

# Compilation of Henry's law constants (version 5.0.0-rc.0) for water as solvent

**Rolf Sander**

Air Chemistry Department, Max-Planck Institute of Chemistry, P.O. Box 3060, 55020 Mainz, Germany

**Correspondence:** Rolf Sander
(rolf.sander@mpic.de)

**Abstract.** Many atmospheric chemicals occur in the gas phase as well as in liquid cloud droplets and aerosol particles. Therefore, it is necessary to understand their distribution between the phases. According to Henry's law, the equilibrium ratio between the abundances in the gas phase and in the aqueous phase is constant for a dilute solution. Henry's law constants of trace gases of potential importance in environmental chemistry have been collected and converted into a uniform format. The compilation contains 46433 values of Henry's law constants for 10173 species, collected from 993 references. It will also be available on the internet at https://www.henrys-law.org. This article is a living review that supersedes the now obsolete publication by Sander (2015).

## 1 Introduction

Henry's law is named after the English chemist William Henry, who studied the topic in the early 19th century. In his publication about the quantity of gases absorbed by water (Henry, 1803), he described the results of his experiments:

> "[...] water takes up, of gas condensed by one, two, or more additional atmospheres, a quantity which, ordinarily compressed, would be equal to twice, thrice, &c. the volume absorbed under the common pressure of the atmosphere."

In other words, the amount of dissolved gas is proportional to its partial pressure in the gas phase. The proportionality factor is called Henry's law constant. In atmospheric chemistry, these constants are needed to describe the distribution of trace species between the air and liquid cloud droplets or aerosol particles. In other areas of environmental research,

the constants are needed to calculate the vaporization of contaminants from rivers and during waste water treatment (e.g. Shen, 1982; Hawthorne et al., 1985; David et al., 2000).

Section 2 provides theoretical background about Henry's law and commonly used quantities and units. In Sect. 3, the compilation of Henry's law constants is described in detail. Additional information can be found in the electronic supplement, which is described in Sect. 4.

This article is a living review describing version 5.0.0-rc.0[1]. Compared to the now obsolete version 4.0 (Sander, 2015), the compilation contains 29083 additional values of Henry's law constants for 5541 additional species, collected from 304 additional references. In cases where experimental data are available for a large temperature range, the data were refitted to a 3-parameter equation, replacing the 2-parameter fits that were used previously. The symbols of the Henry's law constants have been adjusted in order to follow the new recommendations of the International Union of Pure and Applied Chemistry (IUPAC) by Sander et al. (2022). In addition to the CAS registry numbers, chemical species are now also identified by their InChIKeys (Heller et al., 2015).

## 2 Theoretical background

### 2.1 Fundamental types of Henry's law constants

There are many variants of Henry's law constants which can all be classified into two fundamental types: One possibility is to put the aqueous phase into the numerator and the gas phase into the denominator, i.e., define the constant as

---

[1]The name of this version indicates that it is a "release candidate" used for the interactive discussion in ACPD. If necessary, corrections can still be made. It is planned to release the final version 5.0.0 together with the final paper in ACP.



**Table 1.** Variants of Henry's law constants $H$.

| symbol | definition[a] | SI unit | other commonly used (non-SI) units[b] | conversion[c] from $H_\mathrm{s}^{cp}$ |
|---|---|---|---|---|
| Henry's law solubility constants $H_\mathrm{s}$ | | | | |
| $H_\mathrm{s}^{cp}$ | $c_\mathrm{a}/p$ | $\mathrm{mol\,m^{-3}\,Pa^{-1}}$ | M/atm | |
| $H_\mathrm{s}^{xp}$ | $x/p$ | $\mathrm{Pa^{-1}}$ | $\mathrm{atm^{-1}}$ | $H_\mathrm{s}^{xp} = H_\mathrm{s}^{cp} \times M_{\mathrm{H_2O}}/\varrho_{\mathrm{H_2O}}$ |
| $H_\mathrm{s}^{bp}$ | $b/p$ | $\mathrm{mol\,kg^{-1}\,Pa^{-1}}$ | $\mathrm{mol\,kg^{-1}\,atm^{-1}}$ | $H_\mathrm{s}^{bp} = H_\mathrm{s}^{cp}/\varrho_{\mathrm{H_2O}}$ |
| $H_\mathrm{s}^{cc}$ | $c_\mathrm{a}/c_\mathrm{g}$ | 1 (dimensionless) | | $H_\mathrm{s}^{cc} = H_\mathrm{s}^{cp} \times RT$ |
| Henry's law volatility constants $H_\mathrm{v}$ | | | | |
| $H_\mathrm{v}^{pc}$ | $p/c_\mathrm{a}$ | $\mathrm{Pa\,m^3\,mol^{-1}}$ | $\mathrm{atm\,m^3\,mol^{-1}}$ | $H_\mathrm{v}^{pc} = 1/H_\mathrm{s}^{cp}$ |
| $H_\mathrm{v}^{px}$ | $p/x$ | Pa | atm | $H_\mathrm{v}^{px} = (\varrho_{\mathrm{H_2O}}/M_{\mathrm{H_2O}})/H_\mathrm{s}^{cp}$ |
| $H_\mathrm{v}^{pw}$ | $p/w$ | Pa | atm | $H_\mathrm{v}^{pw} = (\varrho_{\mathrm{H_2O}}/M_B)/H_\mathrm{s}^{cp}$ |
| $H_\mathrm{v}^{cc}$ | $c_\mathrm{g}/c_\mathrm{a}$ | 1 (dimensionless) | | $H_\mathrm{v}^{cc} = 1/(H_\mathrm{s}^{cp} \times RT)$ |

[a] The definitions apply only at equilibrium and in the limit of infinite dilution.

[b] Here, M = mol/L and atm = 101 325 Pa.

[c] Here, $M_{\mathrm{H_2O}}$ and $\varrho_{\mathrm{H_2O}}$ are the molar mass and density of water, respectively. $M_B$ is the molar mass of the solute. The simplified conversion formulas are valid only for binary solutions and ideal gases. More conversion formulas can be found in Tab. 2 of Sander et al. (2022).

the quotient $A/G$. Here, $A$ and $G$ are quantities describing the equilibrium composition (at infinite dilution) of the aqueous phase and the gas phase, respectively. Alternatively, the Henry's law constant can be defined as the quotient $G/A$, which results in the inverse value. There is no advantage or disadvantage in using one or the other, the two types exist purely for historical reasons. Unfortunately, the name "Henry's law constant" is used for both types. Therefore, expressions like "a large Henry's law constant" are meaningless unless the type is specified. Especially the dimensionless constants are very error-prone because their type cannot be deduced from the unit. In order to have a consistent terminology, the name "Henry's law solubility constant" (or "Henry solubility" for conciseness) should be used when refering to $A/G$. When refering to $G/A$, the name "Henry's law volatility constant" (or "Henry volatility") should be used.

### 2.2 Variants of Henry's law constants

For both of the fundamental types described in the previous section, there are several variants. This results from the multiplicity of quantities that can be chosen to describe the composition of the two phases. Typical choices for the aqueous phase are molar concentration ($c_\mathrm{a}$), molality ($b$), and amount fraction ($x$). For the gas phase, molar concentration ($c_\mathrm{g}$) and partial pressure ($p$) are often used. Note, however, that it is not possible to use the gas-phase amount fraction ($y$). At a given gas-phase amount fraction, the aqueous-phase concentration $c_\mathrm{a}$ depends on the total pressure and thus the ratio $y/c_\mathrm{a}$ is not a constant.

There are numerous combinations of these quantities. The eight variants recommended by IUPAC are summarized in Table 1. Numerical values of conversion factors between them are shown in Tables 2, 3, and 4.

### 2.3 Symbols

In the current literature, a plethora of different symbols is used for the Henry's law constants. Several symbols are used for the same variant, and sometimes the same symbol is used for different variants. However, for this work a consistent terminology is indispensable. Here, the IUPAC recommendations by Sander et al. (2022) are used: For Henry's law solubility constants, the symbol $H_\mathrm{s}$ is used, and for Henry's law volatility constants the symbol $H_\mathrm{v}$ is used.

To specify the exact variant of the Henry's law constant, two superscripts are used. They refer to the numerator and the denominator of the definition. For example, $H_\mathrm{s}^{cp}$ refers to the Henry solubility defined as $c/p$. If $H_\mathrm{s}$ refers to the reference temperature $T^\ominus = 298.15\,\mathrm{K}$, it will be denoted as $H_\mathrm{s}^\ominus$. A summary of the symbols is shown in Table 5.

### 2.4 Temperature dependence of Henry's law constants

In spite of the name Henry's law "*constant*", it should be kept in mind that its value still depends on some parameters, e.g., the temperature $T$. The temperature dependence of equilibrium constants can be described with the van't Hoff equation, which also applies to Henry's law:

$$\frac{\mathrm{d}\ln H_\mathrm{s}}{\mathrm{d}(1/T)} = \frac{-\Delta_{\mathrm{sol}}H}{R}, \qquad (1)$$

where $\Delta_{\mathrm{sol}}H$ = enthalpy of dissolution and $R$ = gas constant. Note that the letter $H$ in the symbol $\Delta_{\mathrm{sol}}H$ refers to enthalpy and is not related to the letter $H$ for Henry's law constants. Integrating the above equation leads to

$$\ln H_\mathrm{s} = \frac{-\Delta_{\mathrm{sol}}H}{R}\frac{1}{T} + const \qquad (2)$$





**Table 2.** Conversion factors between several Henry's law solubility constants $H_s$ (at $T^{\ominus}$ = 298.15 K and $\varrho^{\ominus}$ = 997 kg/m$^3$).

| | $H_s^{cp} = \ldots \frac{\text{mol}}{\text{m}^3\,\text{Pa}}$ | $H_s^{cp} = \ldots \frac{\text{M}}{\text{atm}}$ | $H_s^{cc} = \ldots$ | $H_s^{bp} = \ldots \frac{\text{mol}}{\text{kg}\,\text{Pa}}$ | $H_s^{bp} = \ldots \frac{\text{mol}}{\text{kg}\,\text{atm}}$ | $H_s^{xp} = \ldots \frac{1}{\text{atm}}$ | $\alpha = \ldots$ |
|---|---|---|---|---|---|---|---|
| $H_s^{cp} = 1\,\frac{\text{mol}}{\text{m}^3\,\text{Pa}}$ | 1.00000 | 101.325 | 2478.96 | $1.00301 \times 10^{-3}$ | 101.630 | 1.83089 | 2271.10 |
| $H_s^{cp} = 1\,\frac{\text{M}}{\text{atm}}$ | $9.86923 \times 10^{-3}$ | 1.00000 | 24.4654 | $9.89893 \times 10^{-6}$ | 1.00301 | 0.0180695 | 22.4140 |
| $H_s^{cc} = 1$ | $4.03395 \times 10^{-4}$ | 0.0408740 | 1.00000 | $4.04609 \times 10^{-7}$ | 0.0409970 | $7.38573 \times 10^{-4}$ | 0.916150 |
| $H_s^{bp} = 1\,\frac{\text{mol}}{\text{kg}\,\text{Pa}}$ | 997.000 | $1.01021 \times 10^5$ | $2.47152 \times 10^6$ | 1.00000 | $1.01325 \times 10^5$ | 1825.40 | $2.26428 \times 10^6$ |
| $H_s^{bp} = 1\,\frac{\text{mol}}{\text{kg}\,\text{atm}}$ | $9.83962 \times 10^{-3}$ | 0.997000 | 24.3920 | $9.86923 \times 10^{-6}$ | 1.00000 | 0.0180153 | 22.3467 |
| $H_s^{xp} = 1\,\frac{1}{\text{atm}}$ | 0.546182 | 55.3419 | 1353.96 | $5.47826 \times 10^{-4}$ | 55.5084 | 1.00000 | 1240.43 |
| $\alpha = 1$ | $4.40316 \times 10^{-4}$ | 0.0446150 | 1.09152 | $4.41641 \times 10^{-7}$ | 0.0447493 | $8.06171 \times 10^{-4}$ | 1.00000 |

**Table 3.** Conversion factors between several Henry's law volatility constants $H_v$ (at $T^{\ominus}$ = 298.15 K and $\varrho^{\ominus}$ = 997 kg/m$^3$).

| | $H_v^{px} = \ldots \text{atm}$ | $H_v^{pc} = \ldots \frac{\text{m}^3\,\text{Pa}}{\text{mol}}$ | $H_v^{pc} = \ldots \frac{\text{m}^3\,\text{atm}}{\text{mol}}$ | $H_v^{cc} = \ldots$ |
|---|---|---|---|---|
| $H_v^{px} = 1\,\text{atm}$ | 1.00000 | 1.83089 | $1.80695 \times 10^{-5}$ | $7.38573 \times 10^{-4}$ |
| $H_v^{pc} = 1\,\frac{\text{m}^3\,\text{Pa}}{\text{mol}}$ | 0.546182 | 1.00000 | $9.86923 \times 10^{-6}$ | $4.03395 \times 10^{-4}$ |
| $H_v^{pc} = 1\,\frac{\text{m}^3\,\text{atm}}{\text{mol}}$ | 55341.9 | $1.01325 \times 10^5$ | 1.00000 | 40.8740 |
| $H_v^{cc} = 1$ | 1353.96 | 2478.96 | 0.0244654 | 1.00000 |

**Table 4.** Products of Henry's law solubility constants $H_s$ and Henry's law volatility constants $H_v$ (at $T^{\ominus}$ = 298.15 K and $\varrho^{\ominus}$ = 997 kg/m$^3$). For example, if $H_v^{px} = 5$ atm, then $H_s^{bp} \approx 11$ mol/(kg atm) because $5 \times 11 \approx 55.5084$.

| | $\dfrac{H_s^{cp}}{\text{mol/(m}^3\,\text{Pa)}}$ | $\dfrac{H_s^{cp}}{\text{M/atm}}$ | $\dfrac{H_s^{cc}}{1}$ | $\dfrac{H_s^{bp}}{\text{mol/(kg Pa)}}$ | $\dfrac{H_s^{bp}}{\text{mol/(kg atm)}}$ | $\dfrac{H_s^{xp}}{\text{1/atm}}$ | $\dfrac{\alpha}{1}$ |
|---|---|---|---|---|---|---|---|
| $\dfrac{H_v^{px}}{\text{atm}}$ | 0.546182 | 55.3419 | 1353.96 | $5.47826 \times 10^{-4}$ | 55.5084 | 1.00000 | 1240.43 |
| $\dfrac{H_v^{pc}}{\text{m}^3\,\text{Pa/mol}}$ | 1.00000 | 101.325 | 2478.96 | $1.00301 \times 10^{-3}$ | 101.630 | 1.83089 | 2271.10 |
| $\dfrac{H_v^{pc}}{\text{m}^3\,\text{atm/mol}}$ | $9.86923 \times 10^{-6}$ | $1.00000 \times 10^{-3}$ | 0.0244654 | $9.89893 \times 10^{-9}$ | $1.00301 \times 10^{-3}$ | $1.80695 \times 10^{-5}$ | 0.0224140 |
| $\dfrac{H_v^{cc}}{1}$ | $4.03395 \times 10^{-4}$ | 0.0408740 | 1.00000 | $4.04609 \times 10^{-7}$ | 0.0409970 | $7.38573 \times 10^{-4}$ | 0.916150 |





**Table 5.** List of Symbols.

| symbol | quantity | SI unit* |
|---|---|---|
| $\varrho$ | density | $\mathrm{kg\,m^{-3}}$ |
| $A$ | parameter for temperature dependence of $H_{\mathrm{s}}$ | 1 |
| $b$ | molality | $\mathrm{mol\,kg^{-1}}$ |
| $B$ | parameter for temperature dependence of $H_{\mathrm{s}}$ | 1 |
| $C$ | parameter for temperature dependence of $H_{\mathrm{s}}$ | 1 |
| $c_{\mathrm{a}}$ | aqueous-phase concentration | $\mathrm{mol\,m^{-3}}$ |
| $c_{\mathrm{g}}$ | gas-phase concentration | $\mathrm{mol\,m^{-3}}$ |
| $D$ | parameter for temperature dependence of $H_{\mathrm{s}}$ | 1 |
| $\Delta_{\mathrm{sol}}H$ | molar enthalpy of dissolution | $\mathrm{J\,mol^{-1}}$ |
| $H_{\mathrm{s}}$ | Henry solubility (all variants) | miscellaneous |
| $H_{\mathrm{s}}^{\ominus}$ | Henry solubility at the reference temperature $T^{\ominus}$ | miscellaneous |
| $H_{\mathrm{s}}^{bp}$ | Henry solubility (defined as $b/p$) | $\mathrm{mol\,kg^{-1}\,;Pa^{-1}}$ |
| $H_{\mathrm{s}}^{cc}$ | Henry solubility (defined as $c/c$) | 1 |
| $H_{\mathrm{s}}^{cp}$ | Henry solubility (defined as $c/p$) | $\mathrm{mol\,m^{-3}\,Pa^{-1}}$ |
| $H_{\mathrm{s,eff}}$ | effective Henry solubility | miscellaneous |
| $H_{\mathrm{s}}'$ | $H \times K_{\mathrm{A}}$ (for strong acids) | miscellaneous |
| $H_{\mathrm{v}}$ | Henry volatility (all variants) | miscellaneous |
| $H_{\mathrm{v}}^{\ominus}$ | Henry volatility at the reference temperature $T^{\ominus}$ | miscellaneous |
| $H_{\mathrm{v}}^{cc}$ | Henry volatility (defined as $c/c$) | 1 |
| $H_{\mathrm{v}}^{pc}$ | Henry volatility (defined as $p/c$) | $\mathrm{Pa\,m^3\,mol^{-1}}$ |
| $H_{\mathrm{v}}^{px}$ | Henry volatility (defined as $p/x$) | Pa |
| $K_{\mathrm{A}}$ | acid constant | $\mathrm{mol\,m^{-3}}$ |
| $M$ | molar mass | $\mathrm{kg\,mol^{-1}}$ |
| $p$ | partial pressure $= c_{\mathrm{g}}RT$ | Pa |
| $R$ | gas constant | $\mathrm{8.314\,J\,(mol\,K)^{-1}}$ |
| $T$ | temperature | K |
| $T^{\ominus}$ | reference temperature | 298.15 K |
| $w$ | mass fraction in the aqueous phase | $\mathrm{kg\,kg^{-1}}$ (dimensionless) |
| $x$ | amount fraction (molar mixing ratio) in the aqueous phase | $\mathrm{mol\,mol^{-1}}$ (dimensionless) |
| $y$ | amount fraction (molar mixing ratio) in the gas phase | $\mathrm{mol\,mol^{-1}}$ (dimensionless) |

*A unit of "1" denotes a quantity of dimension 1, commonly called "dimensionless quantity".

Calling the constant of integration $A$, and defining the parameter $B = -\Delta_{\mathrm{sol}}H/R$, we get:

$$\ln H_{\mathrm{s}} = A + \frac{B}{T} \qquad (3)$$

or

$$H_{\mathrm{s}} = \exp(A) \times \exp\left(\frac{B}{T}\right) \qquad (4)$$

To determine the parameters $A$ and $B$ experimentally, Henry's law constants are measured at several temperatures, and the method of least squares is used to fit the points to a function. Note that functions (3) and (4) produce slightly different fit parameters because the logarithmic function (3) puts less weight on errors of large Henry's law constants than the linear function (4) does. In this work, linear regression is performed using Eq. (3).

Thermodynamic data are often available at the temperature $T^{\ominus} = 298.15$ K. To present Henry's law constants at $T^{\ominus}$ and also show their temperature dependence, an alternative form of Eq. (4) can be used:

$$H_{\mathrm{s}} = H_{\mathrm{s}}^{\ominus} \times \exp\left(B\left(\frac{1}{T} - \frac{1}{T^{\ominus}}\right)\right). \qquad (5)$$

where $H_{\mathrm{s}}^{\ominus} = \exp(A) \times \exp(B/T^{\ominus})$. The enthalpy of dissolution $\Delta_{\mathrm{sol}}H$ is independent of temperature here:

$$\frac{-\Delta_{\mathrm{sol}}H}{R} = \frac{\mathrm{d}\ln H_{\mathrm{s}}}{\mathrm{d}(1/T)} = B \qquad (6)$$

In this work, the values $H_{\mathrm{s}}^{\ominus}$ and $\mathrm{d}\ln H_{\mathrm{s}}/\mathrm{d}(1/T)$ are tabulated.

A simple equation based on the two parameters $A$ and $B$ is valid only for a limited temperature range, in which the enthalpy of dissolution $\Delta_{\mathrm{sol}}H$ can be considered constant. To accommodate a larger temperature range, a third parameter $C$ is often added:

$$\ln H_{\mathrm{s}} = A + \frac{B}{T} + C \times \ln T. \qquad (7)$$

Here, the fit parameters $A$ and $B$ are different from those calculated for function (3). The enthalpy of dissolution



**Table 6.** Temperature-dependent terms and their analytical derivatives. Here, $C$, $C_1$ and $C_2$ are the empirical fit parameters defining $\ln(H_s)$. See Sect. 2.4 for details.

| $\ln(H_s)$ | $\dfrac{\mathrm{d}\ln H_s}{\mathrm{d}(1/T)}$ |
|---|---|
| $C$ | $0$ |
| $C/T$ | $C$ |
| $CT$ | $-CT^2$ |
| $CT^2$ | $-2CT^3$ |
| $C/T^2$ | $2C/T$ |
| $C/T^3$ | $3C/T^2$ |
| $C\ln(T)$ | $-CT$ |
| $C_1\ln(C_2 T)$ | $-C_1 T$ (independent of $C_2$) |
| $C\log_{10}(T)$ | $-CT/\ln(10)$ |

$\Delta_{\mathrm{sol}}H$ changes linearly with temperature in the 3-parameter fit:

$$\frac{-\Delta_{\mathrm{sol}}H}{R} = \frac{\mathrm{d}\ln H_s}{\mathrm{d}(1/T)} = B - CT \tag{8}$$

To cover an even larger temperature range with an empirical formula, the dependence of $\ln H_s$ on $T$ can be expressed as the sum of several terms. For example, Wilhelm et al. (1977) used the formula:

$$\ln H_s = A + B \times T^{-1} + C \times \ln T + D \times T. \tag{9}$$

The analytical derivative is simply the sum of the derivatives of the individual terms. Using the derivatives from Table 6, the temperature dependence of this expression can be calculated as:

$$\frac{\mathrm{d}\ln H_s}{\mathrm{d}(1/T)} = 0 + B - C \times T - D \times T^2. \tag{10}$$

When reporting Henry's law constants as such a function, it is important to present sufficient significant digits because $H_s$ depends exponentially on the parameters.

Note that the temperature dependences for $H_s^{cp}$ and $H_s^{cc}$ are different since the conversion factor between them includes the temperature:

$$
\begin{aligned}
H_s^{cp} &= H_s^{cc}/(RT) \\
\Leftrightarrow \ln H_s^{cp} &= \ln H_s^{cc} + \ln(1/R) + \ln(1/T) \\
\Rightarrow \frac{\mathrm{d}\ln H_s^{cp}}{\mathrm{d}(1/T)} &= \frac{\mathrm{d}\ln H_s^{cc}}{\mathrm{d}(1/T)} + \frac{\mathrm{d}\ln(1/T)}{\mathrm{d}(1/T)} \\
&= \frac{\mathrm{d}\ln H_s^{cc}}{\mathrm{d}(1/T)} + T. \tag{11}
\end{aligned}
$$

## 2.5 Effective Henry's law solubility constants $H_{\mathrm{s,eff}}$

The Henry's law constants mentioned so far describe the equilibrium between a chemical species in the gas phase and exactly the same species in the aqueous phase. This type is called the "intrinsic" Henry's law constant.

Once transferred into the aqueous phase, some species are involved in fast equilibria. In these cases, an "effective" Henry's law constant $H_{\mathrm{s,eff}}$ can be defined, using a "total concentration" $c_{\mathrm{tot}}$. Depending on the chemical class, there are different ways to define such a total concentration.

### 2.5.1 $H_{\mathrm{s,eff}}$ for aldehydes

Aldehydes can be hydrated, forming geminal diols. For example, methanal (HCHO) is almost completely hydrated in aqueous solution:

$$\mathrm{HCHO} + \mathrm{H_2O} \rightleftharpoons \mathrm{H_2C(OH)_2} \tag{12}$$

The total concentration of dissolved methanal is:

$$c_{\mathrm{tot}} = c(\mathrm{HCHO}) + c(\mathrm{H_2C(OH)_2}) \tag{13}$$

The intrinsic Henry's law solubility constant of HCHO is:

$$H_s = \frac{c(\mathrm{HCHO})}{p(\mathrm{HCHO})} \tag{14}$$

In contrast, the effective Henry's law constant $H_{\mathrm{s,eff}}$ is defined as:

$$H_{\mathrm{s,eff}} = \frac{c_{\mathrm{tot}}}{p(\mathrm{HCHO})} = \frac{c(\mathrm{HCHO}) + c(\mathrm{H_2C(OH)_2})}{p(\mathrm{HCHO})} \tag{15}$$

### 2.5.2 $H_{\mathrm{s,eff}}$ for acids and bases

Acids and bases undergo ionic dissociation upon dissolution, e.g.:

$$\mathrm{HCl} \rightleftharpoons \mathrm{H^+} + \mathrm{Cl^-} \tag{16}$$

Defining the total concentration $c_{\mathrm{tot}}$ as

$$c_{\mathrm{tot}} = c(\mathrm{HCl}) + c(\mathrm{Cl^-}), \tag{17}$$

the effective Henry's law constant is:

$$H_{\mathrm{s,eff}} = \frac{c(\mathrm{HCl}) + c(\mathrm{Cl^-})}{p(\mathrm{HCl})} \tag{18}$$

Considering the acidity constant

$$K_{\mathrm{a}} = \frac{c(\mathrm{H^+})\, c(\mathrm{Cl^-})}{c(\mathrm{HCl})}, \tag{19}$$

the relation between the intrinsic and the effective Henry's law constant for HCl can be written as:

$$H_{\mathrm{s,eff}} = H_s \times \left(1 + \frac{K_{\mathrm{a}}}{c(\mathrm{H^+})}\right) \tag{20}$$

Since the factor on the right-hand side contains $c(\mathrm{H^+})$, the conversion between the intrinsic and the effective Henry's law constant is pH-dependent. Proportionality between $p(\mathrm{HCl})$ and $c_{\mathrm{tot}}$ is restricted to conditions under which





the uptake of gaseous HCl does not affect the acidity of the solution. Thus, effective Henry's law constants of acids and bases are not material constants but depend on the solution pH (Sander et al., 2022).

In order to obtain a pH-independent constant, the product of the intrinsic Henry's law constant $H_s^{cp}$ and the acidity constant $K_a$ is often used for HCl and other strong acids:

$$H_s' = H_s^{cp} \times K_a = \frac{c(\text{H}^+) \times c(\text{Cl}^-)}{p(\text{HCl})} \tag{21}$$

Although $H_s'$ is usually also called a Henry's law constant, it should be noted that it is a different quantity and it has different units than $H_s^{cp}$.

### 2.5.3  $H_{s,\text{eff}}$ for halogens

In the aqueous phase, halogens are in equilibrium with their hypohalous acids, e.g.:

$$\text{Cl}_2 + \text{H}_2\text{O} \rightleftharpoons \text{HOCl} + \text{H}^+ + \text{Cl}^-. \tag{22}$$

The equilibrium constant is:

$$K = \frac{c(\text{HOCl})\, c(\text{H}^+)\, c(\text{Cl}^-)}{c(\text{Cl}_2)} \tag{23}$$

Since the sum of $\text{Cl}_2$ and HOCl is not affected by this equilibrium, a "total chlorine concentration" $c_{\text{tot}}$ can be defined as:

$$c_{\text{tot}} = c(\text{Cl}_2) + c(\text{HOCl}), \tag{24}$$

Using $c_{\text{tot}}$, it is formally possible to define an effective Henry's law constant as:

$$H_{s,\text{eff}} = \frac{c_{\text{tot}}}{p(\text{Cl}_2)} = \frac{c(\text{Cl}_2) + c(\text{HOCl})}{p(\text{Cl}_2)} \tag{25}$$

However, this definition is problematic because it doesn't work at infinite dilution. With decreasing $\text{Cl}_2$ concentration, the equilibrium in Eq. (22) will shift to the right (Le Chatelier's principle), i.e., $c(\text{HOCl}) \gg c(\text{Cl}_2)$, and the effective Henry's law solubility constant goes to infinity (e.g., Fig. 2 in Jones (1911) or Fig. A1 in Lin and Pehkonen (1998)). Therefore, the intrinsic Henry's law constant should be used for halogens, and the term "effective Henry's law constant" should be avoided here.

Instead of extrapolating to infinite dilution, the total chlorine solubility is sometimes reported at the fixed partial pressure of $p(\text{Cl}_2)$ = 101325 Pa. However, even in the vicinity of 101325 Pa, the total chlorine concentration $c_{\text{tot}}$ is not proportional to $p(\text{Cl}_2)$.

In order to convert experimentally determined chlorine solubilities to the intrinsic constant $H_s(\text{Cl}_2)$, additional processes may have to be considered, e.g., aqueous-phase diffusion (Brian et al., 1962; Leist, 1986) and the formation of chlorine hydrates (Adams and Edmonds, 1937; Young, 1983).

### 2.6  Dependence of Henry's law constants on the composition of the solution

Values of Henry's law constants for aqueous solutions depend on the composition of the solution, i.e., on its ionic strength and on dissolved organics. In general, the solubility of a gas decreases with increasing salinity ("salting out"). However, a "salting in" effect has also been observed, e.g., for the effective Henry's law constant of glyoxal (Kampf et al., 2013; Kurtén et al., 2015). The effect can be described with the Sechenov equation (Setschenow, 1889). Note that the scientific transliteration from Cyrillic is "Sechenov" but the original article was written in German and used the German transliteration "Setschenow". There are many alternative ways to define the Sechenov equation, depending on how the aqueous-phase composition is described (based on concentration, molality, or molar fraction) and which variant of the Henry's law constant is used. Describing the solution in terms of molality is preferred because molality is invariant to temperature and to the addition of dry salt to the solution (see Sander (1999) for details). Thus, the Sechenov equation can be written as:

$$\log_{10}\left(\frac{H_{s0}^{bp}}{H_s^{bp}}\right) = K_s \times b(\text{salt}) \tag{26}$$

where $H_{s0}^{bp}$ = Henry's law constant in pure water, $H_s^{bp}$ = Henry's law constant in the salt solution, $K_s$ = molality-based Sechenov constant, and $b(\text{salt})$ = molality of the salt.

Since the atmosphere contains very dilute cloud droplets as well as highly concentrated aerosols, adequate values of Henry's law constants should be used. Unfortunately, Sechenov parameters are unknown for many species.

## 3  Values of Henry's law constants

### 3.1  The data compilation

The compilation of Henry's law constants is presented in the appendix, and it will also be available online at https://www.henrys-law.org. It contains Henry's law constants for inorganic and organic species of potential importance in environmental chemistry. Most data were measured at ambient conditions (around 298 K and 1 atm). Data at high temperatures are excluded or (if possible) extrapolated to $T^{\ominus} = 298.15$ K. The data refer to aqueous solutions; octanol and other solvents are not included. The constants refer to pure water as solvent unless noted otherwise (e.g., sea water).

All Henry's law constants have been converted to a uniform format: $H_s^{cp}$ with the unit $\text{mol}/(\text{m}^3\,\text{Pa})$. In cases where the conversion involves the temperature-dependent density of water, the parameterization by Bettin and Spieweck (1990) was used to calculate $\varrho_{\text{H}_2\text{O}}$ at the temperature $T$.

Inorganic substances are sorted according to the elements they contain. The order chosen is: O, H, N, F, Cl, Br, I,



S, rare gases, others. Compounds with several of these elements are put into the last of the applicable sections. For example, nitryl chloride which contains O, N and Cl, is listed in the Cl section. Carbon-containing compounds (including CO and $CO_2$) are sorted somewhat arbitrarily by increasing chain length and complexity. Hetero atoms (O, N, F, Cl, Br, I, S, P, etc.) are sorted in the same order as for inorganic compounds. The table contains the following groups of species:

The first column of the table shows the systematic name, the chemical formula, trivial names (if any), the CAS registry number (in square brackets), and the InChIKey.

The column labeled "$H_s^{cp}$" contains Henry's law solubility constants at the reference temperature $T^{\ominus} = 298.15$ K. Values are rounded to two significant digits and given in the unit mol/($m^3$ Pa).

The column labeled "$\mathrm{d}\ln H_s/\mathrm{d}(1/T)$" contains the temperature dependence of the Henry solubility as defined in Eq. (5), rounded to two significant digits and given in the unit K. If the term $\Delta_{sol}H$ is temperature-dependent, the value of $\mathrm{d}\ln H_s/\mathrm{d}(1/T)$ is calculated at $T^{\ominus} = 298.15$ K.

For each table entry the column labeled "type" denotes how the Henry's law constant was obtained in the given reference. Literature reviews are usually most reliable, followed





by original publications of experimental determinations of $H_s$. Other data has to be treated more carefully. The types listed here are roughly ordered by decreasing reliability:

**"L"**  The cited paper is a *literature* review.

**"M"**  Original publication of a *measured* value.

**"V"**  *Vapor* pressure of the pure substance divided by aqueous solubility (sometimes called "VP/AS").

**"R"**  The cited paper presents a *recalculation* of previously published material (e.g. extrapolation to a different temperature or concentration range).

**"T"**  *Thermodynamical* calculation ($\Delta_{sol}G = -RT \ln H$, see Sander (1999) for details).

**"X"**  The original paper was not available for this study. The data listed here was found in a secondary source.

**"C"**  The paper is a *citation* of a reference which I could not obtain (personal communication, Ph.D. theses, grey literature).

**"Q"**  The value was calculated with a "*quantitative* structure property relationship" (QSPR) or a similar theoretical method.

**"E"**  The value is an *estimate*. Estimates are listed only if no reliable data are available.

**"?"**  The cited paper doesn't clearly state how the value was obtained.

**"W"**  The value is probably wrong, as explained in the note.

In some cases there might be good agreement between different authors. However, if the original work they refer to is not known one has to be careful when evaluating the reliability. It is possible that they were recalculating data from the same source. The similarity in that case would not be due to independent investigations.

The table entries in the pdf of this document are hyperlinked to endnotes with additional information. In order to avoid spreading of erroneous data, some of these notes identify errors in the original publications. Symbols and acronyms used here refer to those in the original publications.

The CAS numbers in the tables are hyperlinked to the NIST Chemistry WebBook.

## 3.2  Further sources of information

### 3.2.1  Review articles

Several reviews about Henry's law have been published, starting with Markham and Kobe (1941), up to more recent publications such as Wilhelm et al. (1977), Mackay and

Shiu (1981), Staudinger and Roberts (1996), Staudinger and Roberts (2001), Fogg and Sangster (2003), and Burkholder et al. (2019). Practical guidance on the use of Henry's law has been published by Smith and Harvey (2007).

Experimental methods to obtain Henry's law constants as well as indirect (theoretical) methods have been described and compared by several authors. Only a brief summary of some articles is given here. For details, the reader is referred to the original publications:

– Battino and Clever (1966): Miscellaneous methods, partially of historical interest

– Betterton (1992): Head-space method, bubble column method, thermodynamic cycles, calculation from vapor pressure and solubility, linear correlations

– Turner et al. (1996): Static methods, mechanical recirculation methods, separate measurement of solubility and pure species vapor pressure, ebulliometry, perturbation chromatography

– Staudinger and Roberts (1996): Batch air stripping, concurrent flow technique, Equilibrium Partitioning in Closed Systems (EPICS), calculation via Quantitative Property Property Relationships (QPPR), Quantitative Structure Property Relationships (QSPR), UNIversal quasichemical Functional group Activity Coefficients (UNIFAC)

– Brennan et al. (1998): Comparison of predictive methods

– Sander (1999): QPPR, QSPR, thermodynamic calculations

– Fogg and Sangster (2003): Miscellaneous methods

– Dupeux et al. (2022): QSPR

### 3.2.2  Internet

On the internet, several pages provide Henry's law constants, e.g.:

– The PubChem database:
https://pubchem.ncbi.nlm.nih.gov

– The NIST Chemistry WebBook:
https://webbook.nist.gov/chemistry

– The ChemSpider database:
https://www.chemspider.com

– The Generator for Explicit Chemistry and Kinetics of Organics in the Atmosphere (GECKO-A) provides Henry's law constants on the basis of experimental data and structure-activity relationships:
http://geckoa.lisa.u-pec.fr/generateur_form.php



- The Pesticide Properties Database (PPD): https://www.ars.usda.gov/Services/docs.htm?docid=14199

- HENRYWIN, a program to calculate Henry's law constants: https://www.epa.gov/tsca-screening-tools/epi-suitetm-estimation-program-interface

- Vapor-liquid equilibrium data (mostly at elevated temperatures) from the "Dortmund Data Bank": http://www.ddbst.com/en/EED/VLE/VLEindex.php

- EPI Suite estimates from the Arctic Monitoring and Assessment Programme (AMAP) database: https://chemicals.amap.no/about

### 3.2.3 Vapor–liquid equilibrium data

Henry's law constants can be obtained from vapor–liquid equilibrium (VLE) data. For example, consider a binary mixture that consists of a solute dissolved in water. The total pressure $P$ over the solution is the sum of the partial pressures of the components. The partial pressure of the solute can be defined via Henry's law, and the partial pressure of the water can be defined via Raoult's law:

$$
\begin{aligned}
P &= p_{\text{solute}} + p_{\text{water}} \\
&= x H_{\text{v}}^{px} + (1-x) p_{\text{sat}} \\
&= x(H_{\text{v}}^{px} - p_{\text{sat}}) + p_{\text{sat}}
\end{aligned} \tag{27}
$$

where $p_{\text{sat}}$ is the saturation vapor pressure of water. If VLE data with the total pressure at several small solute fractions $x$ are available, the derivative $\mathrm{d}P/\mathrm{d}x$ (i.e., the slope of a plot $P$ vs $x$) can be used to obtain the Henry's law constant:

$$
\frac{\mathrm{d}P}{\mathrm{d}x} = H_{\text{v}}^{px} - p_{\text{sat}} \Rightarrow H_{\text{v}}^{px} = \frac{\mathrm{d}P}{\mathrm{d}x} + p_{\text{sat}} \tag{28}
$$

### 4 The electronic supplement

The Supplement contains several files with additional information about the compiled Henry's law constants. It includes a README file with a detailed description. Here, only a short summary is given:

- The files henry_*.f90 contain the Fortran 90 code that was used to convert the values from the original publications to a uniform format: $H_{\text{s}}^{cp}$ with the unit $\mathrm{mol}/(\mathrm{m}^3\,\mathrm{Pa})$. The code and the comments in the code can be used to double-check that the conversion was done correctly.

- If the original publications contain measurements at different temperatures, the code often contains all individual data points, not just the regression line that was used to calculate the temperature dependence. In addition, the supplement contains data files

with the temperature-dependent values of $H_{\text{s}}^{cp}$ in output/Tdep_data/*.dat and plots of the data points as well as the regression lines according to Eqs. (5) and (7) in output/gnuplot/Tdep.pdf.

- If the Henry's law constants are needed in electronic form, it is cumbersome to extract them from the pdf of this article. Therefore, the supplement contains the files output/*.f90 with declarations of the Henry's law constants ($H_{\text{s}}^{cp}$, $H_{\text{s}}^{xp}$, $H_{\text{s}}^{bp}$, $H_{\text{s}}^{cc}$, $H_{\text{v}}^{pc}$, $H_{\text{v}}^{px}$, and $H_{\text{v}}^{cc}$) in Fortran 90 syntax.

- For some references, the util/ directory contains python scripts that preprocess input and perform other calculations related to the original data.

### 5 Summary and outlook

An updated and extended version of a compilation of Henry's law constants has been presented. The collection, which will also be available at https://www.henrys-law.org, will be continously maintained, updated and extended in the future. If necessary, errata will also be posted on the web page. In addition to providing a source of information, I hope that this work will help to identify gaps in our current knowledge and stimulate research projects. In particular, it seems that even for some well-known chemicals like HCl, $\mathrm{Br}_2$, and BrCl, there is a large uncertainty in the value of the Henry's law constants. I always welcome information about new measurements of Henry's law constants to be included in the table.

*Supplement.* The supplement related to this article is available online at: https://doi.org/10.5194/acp-0-1-2023-supplement.

*Competing interests.* The author declares no competing interests.

*Acknowledgements.* Compiling this data set would not have been possible without the help I received. For valuable discussions, bug reports, and also for pointing out and sending copies of additional references to me, I would like to thank W. E. Acree, Jr., C. Allen, W. Asman, G. Ayers, S. Balaz, J. Beauchamp, Y. Chen, N. Couffin, P. J. Crutzen, A. De Visscher, S. Gromov, G. Hart, S. R. Heller, M. Hiatt, S. H. Hilal, R. Ingham, H. S. S. Ip, H.-W. Jacobi, S. Lee, N. Lim, P. J. Linstrom, G. Mallard, J. Montgomery, R. M. Moore, M. Mozurkewich, E. O'Hare, J. Perlinger, P. Riveros, E. Saltzman, S. E. Schwartz, J. Staudinger, J. St-Pierre, J. Überfeld, T. J. Wallington, and J. C. Wheeler. The NIST Chemistry WebBook (https://webbook.nist.gov/chemistry) and PubChem (https://pubchem.ncbi.nlm.nih.gov/) were indispensable tools for obtaining InChIKeys and CAS registry numbers. The Open Babel software (https://openbabel.org) was used a lot to convert between SMILES and other chemical formats. The service charges for this



open access publication have been covered by the Max Planck Society.





## Appendix A: Appendix with data tables

### A1 Inorganic species

#### A1.1 Oxygen (O)

Table A1.1: Oxygen (O)

| Substance Formula (Trivial Name) [CAS Registry Number] InChIKey | $H_s^{cp}$ (at $T^\ominus$) $\left[\dfrac{\text{mol}}{\text{m}^3\,\text{Pa}}\right]$ | $\dfrac{\text{d}\ln H_s^{cp}}{\text{d}(1/T)}$ [K] | Reference | Type | Note |
|---|---|---|---|---|---|
| oxygen | $1.3\times10^{-5}$ | 1500 | Burkholder et al. (2019) | L | 1 |
| O$_2$ | $1.3\times10^{-5}$ | 1500 | Burkholder et al. (2015) | L | 1 |
| [7782-44-7] | | | Clever et al. (2014) | L | 2 |
| MYMOFIZGZYHOMD-UHFFFAOYSA-N | $1.2\times10^{-5}$ | 1700 | Warneck and Williams (2012) | L | |
| | $1.3\times10^{-5}$ | 1500 | Sander et al. (2011) | L | 1 |
| | $1.3\times10^{-5}$ | 1500 | Sander et al. (2006) | L | 1 |
| | $1.3\times10^{-5}$ | 1400 | Fernández-Prini et al. (2003) | L | 3 |
| | $1.3\times10^{-5}$ | 1500 | Battino et al. (1983) | L | |
| | $1.3\times10^{-5}$ | 1500 | Battino (1981) | L | 1 |
| | $1.3\times10^{-5}$ | 1500 | Wilhelm et al. (1977) | L | |
| | $1.2\times10^{-5}$ | 1400 | Himmelblau (1960) | L | 1 |
| | $1.2\times10^{-5}$ | 1600 | Millero et al. (2002a) | M | 4, 5 |
| | $1.2\times10^{-5}$ | 1600 | Millero et al. (2002b) | M | 6, 7 |
| | $1.3\times10^{-5}$ | 1500 | Rettich et al. (2000) | M | 8 |
| | $1.3\times10^{-5}$ | 1400 | Sherwood et al. (1991) | M | 9 |
| | $1.3\times10^{-5}$ | 1500 | Rettich et al. (1981) | M | 10 |
| | $1.3\times10^{-5}$ | 1500 | Cosgrove and Walkley (1981) | M | 11 |
| | $1.2\times10^{-5}$ | | da Silva et al. (1980) | M | 12 |
| | $1.3\times10^{-5}$ | 1400 | Cramer (1980) | M | |
| | $1.3\times10^{-5}$ | 1400 | Benson et al. (1979) | M | |
| | $1.4\times10^{-4}$ | | Razumovskii and Zaikov (1971) | M | 13 |
| | $1.1\times10^{-5}$ | | Power and Stegall (1970) | M | 14 |
| | $1.3\times10^{-5}$ | 1500 | Murray and Riley (1969) | M | 15 |
| | $1.2\times10^{-5}$ | 1200 | Shoor et al. (1969) | M | 16 |
| | $1.2\times10^{-5}$ | 1600 | Carpenter (1966) | M | 17 |
| | $1.3\times10^{-5}$ | 1500 | Morrison and Billett (1952) | M | 18 |
| | $1.2\times10^{-5}$ | | Orcutt and Seevers (1937a) | M | |
| | $1.3\times10^{-5}$ | 1500 | Fox (1909) | M | |
| | $1.2\times10^{-5}$ | 1700 | Geffcken (1904) | M | |
| | $1.3\times10^{-5}$ | 1400 | Winkler (1891b) | M | 19 |
| | $1.3\times10^{-5}$ | 1400 | Bohr and Bock (1891) | M | |
| | $1.2\times10^{-5}$ | 1800 | Timofejew (1890) | M | |
| | $1.2\times10^{-5}$ | 1200 | Bunsen (1855a) | M | |
| | $1.2\times10^{-5}$ | 1600 | Wauchope and Haque (1972) | V | |
| | $1.2\times10^{-5}$ | 1600 | Wauchope and Haque (1972) | V | |
| | $1.3\times10^{-5}$ | 1500 | Wauchope and Haque (1972) | V | |
| | $1.3\times10^{-5}$ | | Pierotti (1965) | T | |
| | $1.4\times10^{-5}$ | | Nunn (1958) | C | 12 |
| | $7.9\times10^{-6}$ | | Hayer et al. (2022) | Q | 20 |
| | $1.3\times10^{-5}$ | 1500 | Yaws et al. (1999) | ? | 21 |
| | $1.1\times10^{-5}$ | | Abraham and Weathersby (1994) | ? | 21 |



Table A1.1: Oxygen (O) (... continued)

| Substance<br>Formula<br>(Trivial Name)<br>[CAS Registry Number]<br>InChIKey | $H_s^{cp}$<br>(at $T^{\ominus}$)<br>$\left[\dfrac{\mathrm{mol}}{\mathrm{m^3\,Pa}}\right]$ | $\dfrac{\mathrm{d}\ln H_s^{cp}}{\mathrm{d}(1/T)}$<br><br>[K] | Reference | Type | Note |
|---|---|---|---|---|---|
| | $1.2\times10^{-5}$ | 1500 | Dean and Lange (1999) | ? | 22, 23 |
| | $1.3\times10^{-5}$ | | Seinfeld (1986) | ? | 21 |
| | $1.3\times10^{-4}$ | | Smith and Bomberger (1980) | ? | 24 |
| ozone | $1.0\times10^{-4}$ | 2800 | Burkholder et al. (2019) | L | |
| $O_3$ | $1.0\times10^{-4}$ | 2800 | Burkholder et al. (2015) | L | |
| [10028-15-6] | | | Clever et al. (2014) | L | 25 |
| CBENFWSGALASAD-UHFFFAOYSA-N | $1.0\times10^{-4}$ | 2800 | Sander et al. (2011) | L | |
| | $1.0\times10^{-4}$ | 2800 | Sander et al. (2006) | L | |
| | $7.6\times10^{-5}$ | 3700 | Biń (2005) | L | |
| | $1.1\times10^{-4}$ | 2400 | Warneck (2003) | L | |
| | | | Battino (1981) | L | 26 |
| | $1.3\times10^{-4}$ | 2000 | Wilhelm et al. (1977) | L | |
| | $1.1\times10^{-4}$ | | Levanov et al. (2008) | M | 12 |
| | $1.1\times10^{-4}$ | 2300 | Gershenzon et al. (2001) | M | |
| | $9.9\times10^{-5}$ | 2600 | Rischbieter et al. (2000) | M | |
| | $9.2\times10^{-5}$ | 2600 | Andreozzi et al. (1996) | M | |
| | $1.2\times10^{-4}$ | 1400 | Sotelo et al. (1989) | M | |
| | $1.1\times10^{-4}$ | 2300 | Kosak-Channing and Helz (1983) | M | |
| | $1.7\times10^{-4}$ | | Gurol and Singer (1982) | M | 12 |
| | | | Roth and Sullivan (1981) | M | 27 |
| | $1.2\times10^{-4}$ | 1900 | Stumm (1958) | M | |
| | $9.9\times10^{-5}$ | 2600 | Kilpatrick et al. (1956) | M | |
| | $1.3\times10^{-4}$ | 2000 | Briner and Perrottet (1939) | M | |
| | $2.0\times10^{-4}$ | | Fischer and Tropsch (1917) | M | |
| | $8.0\times10^{-5}$ | 2900 | Luther (1905) | M | |
| | $1.2\times10^{-4}$ | 4100 | Mailfert (1894) | M | |
| | $1.2\times10^{-4}$ | | Schöne (1873) | M | 28 |
| | $1.1\times10^{-4}$ | 2600 | Chameides (1984) | T | |
| | $1.2\times10^{-4}$ | | Perry and Chilton (1973) | X | 29 |
| | $1.2\times10^{-4}$ | | Hayer et al. (2022) | Q | 20 |
| | | | Lide and Frederikse (1995) | ? | 30 |
| | $9.3\times10^{-5}$ | 2500 | Seinfeld (1986) | ? | 21 |
| | $9.3\times10^{-5}$ | 2500 | Hoffmann and Jacob (1984) | ? | 21 |



### A1.2 Hydrogen (H)

Table A1.2: Hydrogen (H)

| Substance Formula (Trivial Name) [CAS Registry Number] InChIKey | $H_s^{cp}$ (at $T^{\ominus}$) $\left[\dfrac{\mathrm{mol}}{\mathrm{m^3\,Pa}}\right]$ | $\dfrac{\mathrm{d}\ln H_s^{cp}}{\mathrm{d}(1/T)}$ [K] | Reference | Type | Note |
|---|---|---|---|---|---|
| hydrogen atom | $2.6\times10^{-6}$ | | Burkholder et al. (2019) | L | |
| H | $2.6\times10^{-6}$ | | Burkholder et al. (2015) | L | |
| [12385-13-6] | $2.6\times10^{-6}$ | | Sander et al. (2011) | L | |
| YZCKVEUIGOORGS-UHFFFAOYSA-N | $2.6\times10^{-6}$ | | Sander et al. (2006) | L | |
| | $3.1\times10^{-6}$ | | Armstrong et al. (2015) | T | |
| | $3.4\times10^{-6}$ | | Parker (1992) | E | 31 |
| | | | Roduner and Bartels (1992) | ? | 32 |
| hydrogen | $7.8\times10^{-6}$ | 530 | Fernández-Prini et al. (2003) | L | 3 |
| H$_2$ | $7.7\times10^{-6}$ | 490 | Young (1981a) | L | 1 |
| [1333-74-0] | $7.7\times10^{-6}$ | 490 | Wilhelm et al. (1977) | L | |
| UFHFLCQGNIYNRP-UHFFFAOYSA-N | $7.8\times10^{-6}$ | 600 | Himmelblau (1960) | L | 1 |
| | $7.8\times10^{-6}$ | 620 | Schmidt (1979) | M | 33, 34 |
| | $7.7\times10^{-6}$ | 480 | Gordon et al. (1977) | M | 35 |
| | $7.7\times10^{-6}$ | 520 | Crozier and Yamamoto (1974) | M | 36 |
| | $7.2\times10^{-6}$ | | Longo et al. (1970) | M | 14 |
| | $7.2\times10^{-6}$ | | Power and Stegall (1970) | M | 14 |
| | | | Shoor et al. (1969) | M | 37 |
| | $7.5\times10^{-6}$ | | Ruetschi and Amlie (1966) | M | 38 |
| | $7.8\times10^{-6}$ | 510 | Morrison and Billett (1952) | M | 39 |
| | $7.8\times10^{-6}$ | 540 | Geffcken (1904) | M | |
| | $7.7\times10^{-6}$ | 1500 | Braun (1900) | M | 40 |
| | $7.7\times10^{-6}$ | 500 | Winkler (1891a) | M | 41 |
| | $7.5\times10^{-6}$ | 550 | Bohr and Bock (1891) | M | |
| | $7.8\times10^{-6}$ | 610 | Timofejew (1890) | M | |
| | $8.5\times10^{-6}$ | 20 | Bunsen (1855a) | M | 42, 43 |
| | $7.8\times10^{-6}$ | 500 | Wauchope and Haque (1972) | V | |
| | $7.7\times10^{-6}$ | | Hine and Weimar (1965) | R | |
| | $8.3\times10^{-6}$ | | Pierotti (1965) | T | |
| | $6.4\times10^{-6}$ | | Hayer et al. (2022) | Q | 20 |
| | $7.7\times10^{-6}$ | 490 | Yaws et al. (1999) | ? | 21 |
| | $7.7\times10^{-6}$ | | Abraham and Weathersby (1994) | ? | 21 |
| | $7.7\times10^{-6}$ | 500 | Dean and Lange (1999) | ? | 44, 23 |
| deuterium | | | Young (1981a) | L | 45 |
| D$_2$ | $7.9\times10^{-6}$ | 720 | Muccitelli and Wen (1978) | M | 46 |
| [7782-39-0] | $7.9\times10^{-6}$ | 720 | Muccitelli and Wen (1978) | M | |
| UFHFLCQGNIYNRP-VVKOMZTBSA-N | $8.2\times10^{-6}$ | | Hayer et al. (2022) | Q | 20 |
| hydroxyl radical | $3.8\times10^{-1}$ | | Burkholder et al. (2019) | L | |
| OH | $3.8\times10^{-1}$ | | Burkholder et al. (2015) | L | |
| [3352-57-6] | $3.8\times10^{-1}$ | | Sander et al. (2011) | L | |
| TUJKJAMUKRIRHC-UHFFFAOYSA-N | $3.8\times10^{-1}$ | | Sander et al. (2006) | L | |
| | $2.9\times10^{-1}$ | 4300 | Hanson et al. (1992) | T | |
| | $3.2\times10^{-1}$ | | Mozurkewich (1986) | T | |
| | $2.9\times10^{-1}$ | 3100 | Berdnikov and Bazhin (1970) | T | 47 |



Table A1.2: Hydrogen (H) (. . . continued)

| Substance<br>Formula<br>(Trivial Name)<br>[CAS Registry Number]<br>InChIKey | $H_s^{cp}$<br>(at $T^\ominus$)<br>$\left[\dfrac{\mathrm{mol}}{\mathrm{m^3\,Pa}}\right]$ | $\dfrac{\mathrm{d}\ln H_s^{cp}}{\mathrm{d}(1/T)}$<br><br>[K] | Reference | Type | Note |
|---|---|---|---|---|---|
| | $2.5\times10^{-1}$ | | Lelieveld and Crutzen (1991) | C | |
| | $2.0$ | | Lelieveld and Crutzen (1991) | C | |
| | $8.9\times10^{1}$ | | Lelieveld and Crutzen (1991) | C | |
| | $2.5\times10^{-1}$ | 5300 | Jacob (1986) | C | 48 |
| hydroperoxy radical | 6.8 | | Burkholder et al. (2019) | L | |
| HO$_2$ | 6.8 | | Burkholder et al. (2015) | L | |
| [3170-83-0] | 6.8 | | Sander et al. (2011) | L | |
| MHAJPDPJQMAIIY-UHFFFAOYSA-M | 6.8 | | Sander et al. (2006) | L | |
| | $5.7\times10^{1}$ | | Régimbal and Mozurkewich (1997) | R | |
| | $3.8\times10^{1}$ | 5900 | Hanson et al. (1992) | T | |
| | $8.9\times10^{1}$ | | Weinstein-Lloyd and Schwartz (1991) | T | |
| | $8.9\times10^{1}$ | | Chameides (1984) | T | |
| | $1.2\times10^{1}$ | | Schwartz (1984) | T | 49 |
| | $4.6\times10^{1}$ | 4800 | Berdnikov and Bazhin (1970) | T | 47 |
| | | 6600 | Jacob (1986) | E | 50 |
| hydrogen peroxide | $8.6\times10^{2}$ | 7300 | Burkholder et al. (2019) | L | |
| H$_2$O$_2$ | $8.6\times10^{2}$ | 7300 | Burkholder et al. (2015) | L | |
| [7722-84-1] | $9.5\times10^{2}$ | 7200 | Brockbank (2013) | L | 1 |
| MHAJPDPJQMAIIY-UHFFFAOYSA-N | $9.1\times10^{2}$ | 6600 | Warneck and Williams (2012) | L | |
| | $8.3\times10^{2}$ | 7600 | Sander et al. (2011) | L | |
| | $7.6\times10^{2}$ | 7300 | Sander et al. (2006) | L | |
| | $9.8\times10^{2}$ | 6100 | Fogg and Sangster (2003) | L | 51 |
| | $1.2\times10^{3}$ | 5900 | Rivera-Rios (2018) | M | |
| | $1.1\times10^{3}$ | 7000 | Huang and Chen (2010) | M | |
| | $8.2\times10^{2}$ | 7400 | O'Sullivan et al. (1996) | M | |
| | $9.9\times10^{2}$ | 6300 | Lind and Kok (1994) | M | 52 |
| | | | Staffelbach and Kok (1993) | M | 53 |
| | $8.5\times10^{2}$ | 6500 | Zhou and Lee (1992) | M | |
| | $6.7\times10^{2}$ | 7900 | Hwang and Dasgupta (1985) | M | |
| | $1.4\times10^{3}$ | | Yoshizumi et al. (1984) | M | 12 |
| | $9.6\times10^{2}$ | 6600 | Chameides (1984) | T | |
| | $7.0\times10^{2}$ | 7000 | Martin and Damschen (1981) | T | |
| | $6.4\times10^{1}$ | | Hilal et al. (2008) | Q | |
| | $7.0\times10^{2}$ | 7300 | Seinfeld (1986) | ? | 21 |
| | $7.0\times10^{2}$ | 7300 | Hoffmann and Jacob (1984) | ? | 21 |
| | | | Pandis and Seinfeld (1989) | W | 54 |



### A1.3 Nitrogen (N)

Table A1.3: Nitrogen (N)

| Substance Formula (Trivial Name) [CAS Registry Number] InChIKey | $H_s^{cp}$ (at $T^{\ominus}$) $\left[\dfrac{\text{mol}}{\text{m}^3\,\text{Pa}}\right]$ | $\dfrac{\text{d}\ln H_s^{cp}}{\text{d}(1/T)}$ [K] | Reference | Type | Note |
|---|---|---|---|---|---|
| nitrogen | $6.4\times10^{-6}$ | 1300 | Burkholder et al. (2019) | L | 1 |
| $N_2$ | $6.4\times10^{-6}$ | 1300 | Burkholder et al. (2015) | L | 1 |
| [7727-37-9] | $6.4\times10^{-6}$ | 1600 | Warneck and Williams (2012) | L | |
| IJGRMHOSHXDMSA-UHFFFAOYSA-N | $6.4\times10^{-6}$ | 1300 | Sander et al. (2011) | L | 1 |
| | $6.4\times10^{-6}$ | 1300 | Sander et al. (2006) | L | 1 |
| | $6.5\times10^{-6}$ | 1200 | Fernández-Prini et al. (2003) | L | 3 |
| | $6.5\times10^{-6}$ | 1200 | Battino et al. (1984) | L | |
| | $6.5\times10^{-6}$ | 1200 | Battino (1982) | L | 1 |
| | $6.4\times10^{-6}$ | 1300 | Wilhelm et al. (1977) | L | |
| | $5.4\times10^{-6}$ | | Steward et al. (1973) | L | 14 |
| | $6.5\times10^{-6}$ | 1400 | Allott et al. (1973) | L | |
| | $6.3\times10^{-6}$ | 1300 | Himmelblau (1960) | L | 1 |
| | $6.4\times10^{-6}$ | 1300 | Rettich et al. (1984) | M | 55 |
| | $6.6\times10^{-6}$ | 1300 | Cosgrove and Walkley (1981) | M | 11 |
| | $5.5\times10^{-6}$ | | Power and Stegall (1970) | M | 14 |
| | $6.4\times10^{-6}$ | 1300 | Murray et al. (1969) | M | 56 |
| | $6.5\times10^{-6}$ | 1400 | Morrison and Billett (1952) | M | 57 |
| | $6.6\times10^{-6}$ | | Orcutt and Seevers (1937a) | M | |
| | $6.5\times10^{-6}$ | 1100 | Van Slyke et al. (1934) | M | |
| | $5.6\times10^{-6}$ | | Grollman (1929) | M | 58 |
| | $6.4\times10^{-6}$ | 1200 | Fox (1909) | M | |
| | $6.3\times10^{-6}$ | 2200 | Braun (1900) | M | 59 |
| | $6.3\times10^{-6}$ | 1300 | Winkler (1891b) | M | 60 |
| | $6.7\times10^{-6}$ | 1400 | Bohr and Bock (1891) | M | |
| | $5.8\times10^{-6}$ | 1200 | Bunsen (1855a) | M | 43 |
| | $6.4\times10^{-6}$ | 1300 | Wauchope and Haque (1972) | V | |
| | $6.5\times10^{-6}$ | 1300 | Wauchope and Haque (1972) | V | |
| | $6.5\times10^{-6}$ | | Pierotti (1965) | T | |
| | $7.2\times10^{-6}$ | | Nunn (1958) | C | 12 |
| | $5.6\times10^{-6}$ | | Hayer et al. (2022) | Q | 20 |
| | $6.4\times10^{-6}$ | 1600 | Battino et al. (2018) | ? | |
| | $6.3\times10^{-6}$ | 1200 | Yaws et al. (1999) | ? | 21 |
| | $5.7\times10^{-6}$ | | Abraham and Weathersby (1994) | ? | 21 |
| | $6.3\times10^{-6}$ | 1300 | Dean and Lange (1999) | ? | 61, 23 |
| ammonia | $5.9\times10^{-1}$ | 4200 | Burkholder et al. (2019) | L | |
| $NH_3$ | $5.9\times10^{-1}$ | 4200 | Burkholder et al. (2015) | L | |
| [7664-41-7] | $5.9\times10^{-1}$ | 4200 | Sander et al. (2011) | L | |
| QGZKDVFQNNGYKY-UHFFFAOYSA-N | $5.9\times10^{-1}$ | 4200 | Sander et al. (2006) | L | |
| | $5.8\times10^{-1}$ | 4400 | Yoo et al. (1986) | L | 1 |
| | $6.0\times10^{-1}$ | 4200 | Edwards et al. (1978) | L | 1 |
| | $1.0\times10^{-1}$ | 1500 | Wilhelm et al. (1977) | L | |
| | $2.8\times10^{-1}$ | 3200 | Shi et al. (1999) | M | |
| | 9.9 | 6600 | Tsuji et al. (1990) | M | 62 |
| | $6.0\times10^{-1}$ | 4200 | Clegg and Brimblecombe (1989) | M | |



Table A1.3: Nitrogen (N) (...continued)

| Substance<br>Formula<br>(Trivial Name)<br>[CAS Registry Number]<br>InChIKey | $H_s^{cp}$<br>(at $T^{\ominus}$)<br>$\left[\dfrac{\text{mol}}{\text{m}^3\,\text{Pa}}\right]$ | $\dfrac{\text{d}\ln H_s^{cp}}{\text{d}(1/T)}$<br><br>[K] | Reference | Type | Note |
|---|---|---|---|---|---|
| | $5.5\times10^{-1}$ | 4100 | Dasgupta and Dong (1986) | M | |
| | $7.7\times10^{-1}$ | | Holzwarth et al. (1984) | M | |
| | $7.4\times10^{-1}$ | 3700 | Hales and Drewes (1979) | M | |
| | $5.6\times10^{-1}$ | 4200 | Dasgupta and Dong (1986) | T | |
| | $5.7\times10^{-1}$ | 4100 | Chameides (1984) | T | |
| | $5.9\times10^{-1}$ | 4100 | Edwards et al. (1975) | T | 1 |
| | $6.1\times10^{-1}$ | | Van Krevelen et al. (1949) | X | 63 |
| | $2.9\times10^{-1}$ | | Hayer et al. (2022) | Q | 20 |
| | $2.7\times10^{-1}$ | 2400 | Dean and Lange (1999) | ? | 64, 23 |
| | $5.7\times10^{-1}$ | | Abraham et al. (1990) | ? | |
| | $6.1\times10^{-1}$ | 4100 | Seinfeld (1986) | ? | 21 |
| | $5.8\times10^{-1}$ | 4100 | Hoffmann and Jacob (1984) | ? | 21 |
| | $5.2\times10^{-1}$ | | Bone et al. (1983) | ? | 65 |
| hydrazoic acid<br>HN$_3$<br>[7782-79-8]<br>JUINSXZKUKVTMD-UHFFFAOYSA-N | $1.2\times10^{-1}$ | 3800 | Sander et al. (2011) | L | 66 |
| | $9.8\times10^{-2}$ | 3100 | Wilhelm et al. (1977) | L | |
| | $1.2\times10^{-1}$ | 3700 | Betterton and Robinson (1997) | M | |
| | $9.9\times10^{-2}$ | | Templeton and King (1971) | M | 38 |
| | $7.6\times10^{-1}$ | | Modarresi et al. (2007) | Q | 67 |
| | | | Burkholder et al. (2019) | W | 68 |
| | | | Burkholder et al. (2015) | W | 69 |
| hydrazine<br>H$_4$N$_2$<br>[302-01-2]<br>OAKJQQAXSVQMHS-UHFFFAOYSA-N | $1.6\times10^{1}$ | | HSDB (2015) | V | |
| dinitrogen monoxide<br>N$_2$O<br>(nitrous oxide; laughing gas)<br>[10024-97-2]<br>GQPLMRYTRLFLPF-UHFFFAOYSA-N | $2.4\times10^{-4}$ | 2600 | Burkholder et al. (2019) | L | 1 |
| | $2.1\times10^{-4}$ | 2600 | Burkholder et al. (2019) | L | 70 |
| | $2.4\times10^{-4}$ | 2600 | Burkholder et al. (2015) | L | 1 |
| | $2.1\times10^{-4}$ | 2600 | Burkholder et al. (2015) | L | 70 |
| | $2.4\times10^{-4}$ | 2700 | Warneck and Williams (2012) | L | |
| | $2.4\times10^{-4}$ | 2600 | Sander et al. (2011) | L | 1 |
| | $2.4\times10^{-4}$ | 2600 | Sander et al. (2006) | L | 1 |
| | $2.4\times10^{-4}$ | 2500 | Young (1981b) | L | 1 |
| | $2.4\times10^{-4}$ | 2600 | Wilhelm et al. (1977) | L | |
| | $1.8\times10^{-4}$ | | Steward et al. (1973) | L | 14 |
| | $2.5\times10^{-4}$ | 2500 | Allott et al. (1973) | L | |
| | $2.4\times10^{-4}$ | 2500 | Weiss and Price (1980) | M | 71 |
| | $2.5\times10^{-4}$ | 2300 | Gabel and Schultz (1973) | M | |
| | $2.4\times10^{-4}$ | | Joosten and Danckwerts (1972) | M | |
| | $1.9\times10^{-4}$ | | Bachofen and Farhi (1971) | M | 14 |
| | $2.4\times10^{-4}$ | 2400 | Saidman et al. (1966) | M | |
| | $1.4\times10^{-4}$ | | Sy and Hasbrouck (1964) | M | 14 |
| | $2.2\times10^{-4}$ | | Nunn (1958) | M | 72 |
| | $2.4\times10^{-4}$ | | Orcutt and Seevers (1937a) | M | |
| | $2.4\times10^{-4}$ | 2500 | Kunerth (1922) | M | |
| | $2.4\times10^{-4}$ | 2400 | Siebeck (1909) | M | |
| | $2.4\times10^{-4}$ | 2700 | Geffcken (1904) | M | |



Table A1.3: Nitrogen (N) (... continued)

| Substance Formula (Trivial Name) [CAS Registry Number] InChIKey | $H_\mathrm{s}^{cp}$ (at $T^\ominus$) $\left[\dfrac{\mathrm{mol}}{\mathrm{m^3\,Pa}}\right]$ | $\dfrac{\mathrm{d}\ln H_\mathrm{s}^{cp}}{\mathrm{d}(1/T)}$ [K] | Reference | Type | Note |
|---|---|---|---|---|---|
| | $2.3\times10^{-4}$ | 2900 | Roth (1897) | M | 73 |
| | $2.5\times10^{-4}$ | 2600 | Carius (1855) | M | |
| | $2.6\times10^{-4}$ | 2500 | Gordon (1895) | X | 74 |
| | $1.7\times10^{-4}$ | | Harris (1951) | X | 14, 75 |
| | $3.0\times10^{-4}$ | | Macintosh et al. (1958) | X | 12, 75 |
| | $1.7\times10^{-4}$ | | Orcutt and Seevers (1937b) | X | 58, 75 |
| | $2.8\times10^{-4}$ | | Nunn (1958) | C | 12 |
| | $2.2\times10^{-4}$ | | Hayer et al. (2022) | Q | 20 |
| | | 3600 | Kühne et al. (2005) | Q | |
| | | 2700 | Kühne et al. (2005) | ? | |
| | $2.4\times10^{-4}$ | 2500 | Yaws et al. (1999) | ? | 21 |
| | $1.8\times10^{-4}$ | | Abraham and Weathersby (1994) | ? | 21 |
| | $2.4\times10^{-4}$ | 2800 | Dean and Lange (1999) | ? | 23 |
| | $2.5\times10^{-4}$ | | Seinfeld (1986) | ? | 21 |
| | $2.5\times10^{-4}$ | | Liss and Slater (1974) | ? | |
| nitrogen monoxide NO (nitric oxide) [10102-43-9] MWUXSHHQAYIFBG-UHFFFAOYSA-N | $1.9\times10^{-5}$ | 1600 | Warneck and Williams (2012) | L | |
| | $1.9\times10^{-5}$ | 1600 | Sander et al. (2011) | L | 76, 1 |
| | $1.9\times10^{-5}$ | 1600 | Sander et al. (2006) | L | 77, 1 |
| | $1.9\times10^{-5}$ | 1500 | Schwartz and White (1981) | L | |
| | $1.9\times10^{-5}$ | 1400 | Young (1981b) | L | 78, 1 |
| | $1.3\times10^{-5}$ | | Zafiriou and McFarland (1980) | M | 79 |
| | $2.3\times10^{-5}$ | | Komiyama and Inoue (1980) | M | 80 |
| | $1.9\times10^{-5}$ | 1500 | Komiyama and Inoue (1978) | M | |
| | $1.9\times10^{-5}$ | 1400 | Winkler (1901) | M | 81 |
| | $3.4\times10^{-5}$ | | Pierotti (1965) | T | |
| | $1.9\times10^{-5}$ | 1300 | Loomis (1928) | C | 82 |
| | $1.5\times10^{-5}$ | | Hayer et al. (2022) | Q | 20 |
| | | 1500 | Kühne et al. (2005) | Q | |
| | | 1600 | Kühne et al. (2005) | ? | |
| | $1.9\times10^{-5}$ | 1400 | Yaws et al. (1999) | ? | 21 |
| | $1.9\times10^{-5}$ | 1500 | Dean and Lange (1999) | ? | 83, 23 |
| | $1.9\times10^{-5}$ | | Seinfeld (1986) | ? | 21 |
| | $1.9\times10^{-5}$ | | Andrew and Hanson (1961) | ? | |
| | | | Burkholder et al. (2019) | W | 84 |
| | | | Burkholder et al. (2015) | W | 85 |
| | | | Wilhelm et al. (1977) | W | 86 |
| nitrogen dioxide NO$_2$ [10102-44-0] JCXJVPUVTGWSNB-UHFFFAOYSA-N | $1.2\times10^{-4}$ | 2400 | Burkholder et al. (2019) | L | |
| | $1.2\times10^{-4}$ | 2400 | Burkholder et al. (2015) | L | |
| | $9.9\times10^{-5}$ | | Warneck and Williams (2012) | L | |
| | $1.2\times10^{-4}$ | 2400 | Sander et al. (2011) | L | |
| | $1.4\times10^{-4}$ | | Sander et al. (2006) | L | |
| | $1.2\times10^{-4}$ | | Schwartz and White (1981) | L | |
| | $1.4\times10^{-4}$ | | Cheung et al. (2000) | M | |
| | $6.9\times10^{-5}$ | | Lee and Schwartz (1981) | M | 87 |
| | $2.3\times10^{-4}$ | | Komiyama and Inoue (1980) | M | 80 |
| | $1.2\times10^{-4}$ | 2500 | Chameides (1984) | T | |



Table A1.3: Nitrogen (N) (...continued)

| Substance Formula (Trivial Name) [CAS Registry Number] InChIKey | $H_s^{cp}$ (at $T^{\ominus}$) $\left[\dfrac{\text{mol}}{\text{m}^3\,\text{Pa}}\right]$ | $\dfrac{\text{d}\ln H_s^{cp}}{\text{d}(1/T)}$ [K] | Reference | Type | Note |
|---|---|---|---|---|---|
|  | $3.4\times10^{-4}$ | 1800 | Berdnikov and Bazhin (1970) | T | 47 |
|  | $9.9\times10^{-5}$ |  | Pandis and Seinfeld (1989) | ? | 88 |
|  | $9.9\times10^{-5}$ |  | Seinfeld (1986) | ? | 21 |
|  | $4.0\times10^{-4}$ |  | Andrew and Hanson (1961) | ? |  |
| nitrogen trioxide $NO_3$ (nitrate radical) [12033-49-7] YPJKMVATUPSWOH-UHFFFAOYSA-N | $3.8\times10^{-4}$ |  | Burkholder et al. (2019) | L |  |
|  | $3.8\times10^{-4}$ |  | Burkholder et al. (2015) | L |  |
|  | $3.8\times10^{-4}$ |  | Sander et al. (2011) | L |  |
|  | $3.8\times10^{-4}$ |  | Sander et al. (2006) | L |  |
|  | $1.8\times10^{-2}$ |  | Thomas et al. (1998) | M |  |
|  | $5.9\times10^{-3}$ |  | Rudich et al. (1996) | M | 89 |
|  | $1.2\times10^{-1}$ | 1900 | Chameides (1986) | T |  |
|  | $3.4\times10^{-4}$ | 2000 | Berdnikov and Bazhin (1970) | T | 47 |
|  |  |  | Jacob (1986) | E | 90 |
|  |  |  | Seinfeld and Pandis (1998) | ? | 91 |
| dinitrogen trioxide $N_2O_3$ [10544-73-7] LZDSILRDTDCIQT-UHFFFAOYSA-N | $5.9\times10^{-3}$ |  | Schwartz and White (1981) | L |  |
|  | $2.5\times10^{-1}$ |  | Komiyama and Inoue (1978) | M |  |
| dinitrogen tetroxide $N_2O_4$ [10544-72-6] WFPZPJSADLPSON-UHFFFAOYSA-N | $1.4\times10^{-2}$ |  | Schwartz and White (1981) | L |  |
|  | $2.0\times10^{-2}$ |  | Komiyama and Inoue (1980) | M | 80 |
|  | $1.6\times10^{-2}$ | 3500 | Komiyama and Inoue (1978) | M |  |
|  | $3.1\times10^{-2}$ |  | Andrew and Hanson (1961) | M |  |
|  | $1.3\times10^{-2}$ | 1100 | Kramers et al. (1961) | M |  |
| dinitrogen pentoxide $N_2O_5$ (nitric anhydride) [10102-03-1] ZWWCURLKEXEFQT-UHFFFAOYSA-N | $3.0\times10^{-2}$ |  | Cruzeiro et al. (2022) | T | 92 |
|  | $3.9\times10^{-3}$ |  | Galib and Limmer (2021) | T | 93 |
|  | $4.9\times10^{-3}$ |  | Hirshberg et al. (2018) | T | 94 |
|  | $8.7\times10^{-4}$ | 3600 | Fried et al. (1994) | T | 95 |
|  | $3.9\times10^{-2}$ | 4300 | Robinson et al. (1997) | Q | 96 |
|  | $4.9\times10^{-2}$ |  | Mentel et al. (1999) | E | 97 |
|  | $\infty$ |  | Sander and Crutzen (1996) | E | 98 |
|  | $\infty$ |  | Jacob (1986) | E | 98 |
| hydroxylamine $H_3NO$ [7803-49-8] AVXURJPOCDRRFD-UHFFFAOYSA-N | $1.4\times10^{3}$ |  | HSDB (2015) | Q | 99 |
| nitrous acid $HNO_2$ [7782-77-6] IOVCWXUNBOPUCH-UHFFFAOYSA-N | $4.8\times10^{-1}$ | 4800 | Schwartz and White (1981) | L |  |
|  | $4.7\times10^{-1}$ | 4900 | Becker et al. (1998) | M |  |
|  | $4.7\times10^{-1}$ | 4900 | Becker et al. (1996) | M |  |
|  | $4.8\times10^{-1}$ | 4900 | Park and Lee (1988) | M |  |
|  | $3.7\times10^{-1}$ | 9000 | Komiyama and Inoue (1978) | M |  |
|  | $4.7\times10^{-1}$ | 4700 | Martin (1984) | T |  |
|  | $4.8\times10^{-1}$ | 4800 | Chameides (1984) | T |  |
|  | $4.8\times10^{-1}$ |  | Seinfeld (1986) | ? | 21 |





Table A1.3: Nitrogen (N) (... continued)

| Substance<br>Formula<br>(Trivial Name)<br>[CAS Registry Number]<br>InChIKey | $H_s^{cp}$<br>(at $T^{\ominus}$)<br>$\left[\dfrac{\text{mol}}{\text{m}^3\,\text{Pa}}\right]$ | $\dfrac{\text{d}\ln H_s^{cp}}{\text{d}(1/T)}$<br><br>[K] | Reference | Type | Note |
|---|---|---|---|---|---|
| nitric acid | $8.8\times10^2$ | | Durham et al. (1981) | V | |
| HNO$_3$ | $2.1\times10^3$ | 8700 | Lelieveld and Crutzen (1991) | R | 100 |
| [7697-37-2] | | | Clegg and Brimblecombe (1990) | T | 101, 1 |
| GRYLNZFGIOXLOG-UHFFFAOYSA-N | | | Brimblecombe and Clegg (1989) | T | 102 |
| | $2.6\times10^4$ | 8700 | Chameides (1984) | T | |
| | $2.1\times10^3$ | | Schwartz and White (1981) | T | |
| | $2.1\times10^3$ | | Pandis and Seinfeld (1989) | ? | 103 |
| | $2.1\times10^3$ | | Seinfeld (1986) | ? | 21 |
| | $3.4\times10^3$ | 8800 | Hoffmann and Jacob (1984) | ? | 21 |
| | | | Brimblecombe and Clegg (1988) | W | 104 |
| pernitric acid | $3.9\times10^{-1}$ | 8400 | Leu and Zhang (1999) | L | |
| HNO$_4$ | $3.9\times10^1$ | | Amels et al. (1996) | M | |
| [26404-66-0] | $1.2\times10^2$ | 6900 | Régimbal and Mozurkewich (1997) | T | |
| UUZZMWZGAZGXSF-UHFFFAOYSA-N | $1.4\times10^2$ | | Warneck (1999) | C | |
| | $2.0\times10^2$ | 0 | Jacob et al. (1989) | C | |
| | | | Möller and Mauersberger (1992) | E | 105 |



### A1.4  Fluorine (F)

Table A1.4: Fluorine (F)

| Substance Formula (Trivial Name) [CAS Registry Number] InChIKey | $H_s^{cp}$ (at $T^{\ominus}$) $\left[\dfrac{\mathrm{mol}}{\mathrm{m^3\,Pa}}\right]$ | $\dfrac{\mathrm{d}\ln H_s^{cp}}{\mathrm{d}(1/T)}$ [K] | Reference | Type | Note |
|---|---|---|---|---|---|
| fluorine atom<br>F<br>[14762-94-8]<br>YCKRFDGAMUMZLT-UHFFFAOYSA-N | $2.0\times10^{-4}$ | 400 | Berdnikov and Bazhin (1970) | T | 47 |
| hydrogen fluoride<br>HF<br>[7664-39-3]<br>KRHYYFGTRYWZRS-UHFFFAOYSA-N | $1.3\times10^{2}$<br><br>$6.8\times10^{-4}$ | | Fredenhagen and Wellmann (1932a)<br>Brimblecombe and Clegg (1989)<br>Hayer et al. (2022)<br>Brimblecombe and Clegg (1988) | M<br>T<br>Q<br>W | <br>106<br>20<br>104 |
| difluorine monoxide<br>$F_2O$<br>[7783-41-7]<br>UJMWVICAENGCRF-UHFFFAOYSA-N | $2.9\times10^{-5}$ | | Schäfer and Lax (1962) | C | |
| nitrogen trifluoride<br>$NF_3$<br>[7783-54-2]<br>GVGCUCJTUSOZKP-UHFFFAOYSA-N | $7.9\times10^{-6}$<br>$7.9\times10^{-6}$<br>$7.9\times10^{-6}$<br>$7.9\times10^{-6}$<br>$7.7\times10^{-6}$<br>$7.8\times10^{-6}$ | 1900<br>1900<br>1900<br>1900<br>1700<br>1900 | Burkholder et al. (2019)<br>Burkholder et al. (2015)<br>Sander et al. (2011)<br>Wilhelm et al. (1977)<br>Dean et al. (1973)<br>Ashton et al. (1968) | L<br>L<br>L<br>L<br>M<br>M | 1<br>1<br>1<br><br>107<br>108 |
| dinitrogen tetrafluoride<br>$N_2F_4$<br>(tetrafluorohydrazine)<br>[10036-47-2]<br>GFADZIUESKAXAK-UHFFFAOYSA-N | $8.4\times10^{-6}$<br>$8.4\times10^{-6}$<br>$8.4\times10^{-6}$<br>$8.4\times10^{-6}$<br>$8.4\times10^{-6}$ | 2500<br>2500<br>2500<br>2500<br>2400 | Burkholder et al. (2019)<br>Burkholder et al. (2015)<br>Sander et al. (2011)<br>Wilhelm et al. (1977)<br>Dean et al. (1973) | L<br>L<br>L<br>L<br>M | 1<br>1<br>1<br><br>109 |





### A1.5 Chlorine (Cl)

Table A1.5: Chlorine (Cl)

| Substance Formula (Trivial Name) [CAS Registry Number] InChIKey | $H_s^{cp}$ (at $T^\ominus$) $\left[\dfrac{\text{mol}}{\text{m}^3\,\text{Pa}}\right]$ | $\dfrac{\text{d}\ln H_s^{cp}}{\text{d}(1/T)}$ [K] | Reference | Type | Note |
|---|---|---|---|---|---|
| chlorine (molecular) | | | Burkholder et al. (2019) | L | 110 |
| Cl$_2$ | | | Burkholder et al. (2015) | L | 110 |
| [7782-50-5] | | | Sander et al. (2011) | L | 110 |
| KZBUYRJDOAKODT-UHFFFAOYSA-N | | | Sander et al. (2006) | L | 110 |
| | | | Young (1983) | L | 111 |
| | $6.1\times10^{-4}$ | 3200 | Aieta and Roberts (1986) | M | |
| | $6.2\times10^{-4}$ | 3500 | Whitney and Vivian (1941a) | M | 112 |
| | | | Jones (1911) | M | 110 |
| | | | Winkler (1907) | M | 110 |
| | $6.0\times10^{-4}$ | 3000 | Yakovkin (1900) | M | 113 |
| | | | Bakhuis Roozeboom (1884) | M | 110 |
| | | | Goodwin (1883) | M | 110 |
| | | | Schoenfeld (1855) | M | 110 |
| | $7.4\times10^{-4}$ | 2600 | Lin and Pehkonen (1998) | R | |
| | $5.9\times10^{-4}$ | | Leaist (1986) | R | 114 |
| | | 3200 | Brian et al. (1962) | R | |
| | | | Adams and Edmonds (1937) | R | 115 |
| | | | Arkadiev (1918) | R | 116 |
| | $6.1\times10^{-4}$ | 2800 | Wagman et al. (1982) | T | |
| | $8.7\times10^{-4}$ | | Hayer et al. (2022) | Q | 20 |
| | | | Bartlett and Margerum (1999) | ? | 110, 21, 117 |
| | | | Yaws et al. (1999) | ? | 110 |
| | | | Dean and Lange (1999) | ? | 110 |
| | | | Wilhelm et al. (1977) | ? | 118, 110 |
| chlorine atom | $2.3\times10^{-2}$ | | Burkholder et al. (2019) | L | |
| Cl | $2.3\times10^{-2}$ | | Burkholder et al. (2015) | L | |
| [22537-15-1] | $2.3\times10^{-2}$ | | Sander et al. (2011) | L | |
| ZAMOUSCENKQFHK-UHFFFAOYSA-N | $2.3\times10^{-2}$ | | Sander et al. (2006) | L | |
| | $2.0\times10^{-3}$ | | Mozurkewich (1986) | T | 119 |
| | $1.5\times10^{-4}$ | 1500 | Berdnikov and Bazhin (1970) | T | 47 |
| hydrogen chloride | | | Clegg and Brimblecombe (1986) | L | 120 |
| HCl | $1.5\times10^{1}$ | | Chen et al. (1979) | R | |
| [7647-01-0] | | | Carslaw et al. (1995) | T | 121, 1 |
| VEXZGXHMUGYJMC-UHFFFAOYSA-N | | | Brimblecombe and Clegg (1989) | T | 122 |
| | $1.1\times10^{-2}$ | 2300 | Marsh and McElroy (1985) | T | |
| | | | Wagman et al. (1982) | T | 123 |
| | $2.0\times10^{-1}$ | | Graedel and Goldberg (1983) | C | |
| | $2.4\times10^{-1}$ | | Hayer et al. (2022) | Q | 20 |
| | | | Seinfeld and Pandis (1998) | ? | 91 |
| | $1.9\times10^{-1}$ | 620 | Dean and Lange (1999) | ? | 124, 23 |
| | $2.5\times10^{1}$ | | Seinfeld (1986) | ? | 21 |
| | 7.2 | 2000 | Pandis and Seinfeld (1989) | W | 125 |
| | | | Brimblecombe and Clegg (1988) | W | 104 |





Table A1.5: Chlorine (Cl) (...continued)

| Substance Formula (Trivial Name) [CAS Registry Number] InChIKey | $H_s^{cp}$ (at $T^{\ominus}$) $\left[\dfrac{\mathrm{mol}}{\mathrm{m}^3\,\mathrm{Pa}}\right]$ | $\dfrac{\mathrm{d}\ln H_s^{cp}}{\mathrm{d}(1/T)}$ [K] | Reference | Type | Note |
|---|---|---|---|---|---|
| hypochlorous acid | 6.5 | 5900 | Burkholder et al. (2019) | L | |
| HOCl | 6.5 | 5900 | Burkholder et al. (2015) | L | |
| [7790-92-3] | 6.5 | 5900 | Sander et al. (2011) | L | |
| QWPPOHNGKGFGJK-UHFFFAOYSA-N | 6.5 | 5900 | Sander et al. (2006) | L | |
| | 6.5 | 5900 | Huthwelker et al. (1995) | L | |
| | 9.1 | | Blatchley et al. (1992) | M | 12 |
| | 4.7 | 1600 | Hanson and Ravishankara (1991) | M | 126 |
| | 9.0 | | McCoy et al. (1990) | M | 12 |
| | 6.0 | 4900 | Holzwarth et al. (1984) | M | 127 |
| | $1.2\times10^1$ | 5200 | Imagawa (1950) | M | 11 |
| | 6.4 | 8900 | Ourisson and Kastner (1939) | M | |
| | 2.6 | 5100 | Wagman et al. (1982) | T | |
| | 5.4 | | Hilal et al. (2008) | Q | |
| perchloric acid | $9.9\times10^3$ | | Jaeglé et al. (1996) | E | 128 |
| HClO$_4$ | | | | | |
| [7601-90-3] | | | | | |
| VLTRZXGMWDSKGL-UHFFFAOYSA-N | | | | | |
| monochlorine monoxide | $7.0\times10^{-3}$ | | Burkholder et al. (2019) | L | |
| ClO | $7.0\times10^{-3}$ | | Burkholder et al. (2015) | L | |
| [14989-30-1] | $7.0\times10^{-3}$ | | Sander et al. (2011) | L | |
| MLWGAEVSWJXOQJ-UHFFFAOYSA-N | $7.0\times10^{-3}$ | | Sander et al. (2006) | L | |
| dichlorine monoxide | | | Secoy and Cady (1941) | M | 129 |
| Cl$_2$O | | | Ourisson and Kastner (1939) | M | 130 |
| [7791-21-1] | $3.4\times10^{-2}$ | 5900 | this work | R | 131, 132, 133 |
| RCJVRSBWZCNNQT-UHFFFAOYSA-N | $7.6\times10^{-2}$ | 5600 | this work | R | 132, 134 |
| | $7.0\times10^{-2}$ | | Roth (1942) | R | 135 |
| | | | Burkholder et al. (2019) | ? | 136 |
| | | | Burkholder et al. (2015) | ? | 136 |
| | | | Sander et al. (2011) | ? | 136 |
| | | | Sander et al. (2006) | ? | 136 |
| | | | Young (1983) | ? | 137 |
| | $1.7\times10^{-1}$ | 1800 | Wilhelm et al. (1977) | ? | 138 |
| chlorine dioxide | $1.0\times10^{-2}$ | 3500 | Burkholder et al. (2019) | L | |
| ClO$_2$ | $1.0\times10^{-2}$ | 3500 | Burkholder et al. (2015) | L | |
| [10049-04-4] | $1.0\times10^{-2}$ | 3500 | Sander et al. (2011) | L | |
| OSVXSBDYLRYLIG-UHFFFAOYSA-N | $1.0\times10^{-2}$ | 3500 | Sander et al. (2006) | L | |
| | $1.0\times10^{-2}$ | 3300 | Young (1983) | L | 1 |
| | $1.0\times10^{-2}$ | 3300 | Wilhelm et al. (1977) | L | |
| | $1.0\times10^{-2}$ | 3200 | Kepinski and Trzeszczynski (1964) | M | |
| | $1.1\times10^{-2}$ | 3100 | Ishi (1958) | M | |
| | $1.0\times10^{-2}$ | | Taube and Dodgen (1949) | M | |
| | $6.6\times10^{-3}$ | 1200 | Bigorgne (1947) | M | |
| | $9.7\times10^{-3}$ | 3600 | Holst (1944) | M | |
| | | | Haller and Northgraves (1955) | C | 139 |



Table A1.5: Chlorine (Cl) (... continued)

| Substance<br>Formula<br>(Trivial Name)<br>[CAS Registry Number]<br>InChIKey | $H_\mathrm{s}^{cp}$<br>(at $T^\ominus$)<br>$\left[\dfrac{\mathrm{mol}}{\mathrm{m^3\,Pa}}\right]$ | $\dfrac{\mathrm{d}\ln H_\mathrm{s}^{cp}}{\mathrm{d}(1/T)}$<br><br>[K] | Reference | Type | Note |
|---|---|---|---|---|---|
| | $1.1\times10^{-2}$ | 3100 | Mavu (2011) | ? | |
| | $9.7\times10^{-3}$ | | Morrow et al. (2006) | ? | |
| | $1.0\times10^{-2}$ | 3300 | Yaws et al. (1999) | ? | 21 |
| perchloryl fluoride<br>ClO$_3$F<br>[7616-94-6]<br>XHFXMNZYIKFCPN-UHFFFAOYSA-N | $4.4\times10^{-5}$ | | Hayer et al. (2022) | Q | 20 |
| nitrosyl chloride<br>NOCl<br>[2696-92-6]<br>VPCDQGACGWYTMC-UHFFFAOYSA-N | $>4.9\times10^{-4}$<br>$>4.9\times10^{-4}$<br>$>4.9\times10^{-4}$ | | Burkholder et al. (2019)<br>Burkholder et al. (2015)<br>Scheer et al. (1997) | L<br>L<br>M | |
| nitryl chloride<br>ClNO$_2$<br>[13444-90-1]<br>HSSFHZJIMRUXDM-UHFFFAOYSA-M | $4.5\times10^{-4}$<br>$2.4\times10^{-4}$<br>$3.9\times10^{-4}$ | | Frenzel et al. (1998)<br>Behnke et al. (1997)<br>Roberts et al. (2008) | E<br>E<br>? | <br>140<br> |
| chlorine nitrate<br>ClNO$_3$<br>[14545-72-3]<br>XYLGPCWDPLOBGP-UHFFFAOYSA-N | $7.0\times10^{-2}$<br>$\infty$ | 4500 | Robinson et al. (1997)<br>Sander and Crutzen (1996) | Q<br>E | 141<br>98 |
| chloramine<br>NH$_2$Cl<br>(chloramide)<br>[10599-90-3]<br>QDHHCQZDFGDHMP-UHFFFAOYSA-N | $8.6\times10^{-1}$<br>$8.6\times10^{-1}$<br>$8.6\times10^{-1}$<br>$8.6\times10^{-1}$<br>$9.2\times10^{-1}$<br>$4.6\times10^{-5}$ | 6000<br>6000<br>6000<br>6000<br>4800<br> | Burkholder et al. (2019)<br>Burkholder et al. (2015)<br>Sander et al. (2011)<br>Sander et al. (2006)<br>Holzwarth et al. (1984)<br>Hayer et al. (2022) | L<br>L<br>L<br>L<br>M<br>Q | <br><br><br><br><br>20 |
| dichloramine<br>NHCl$_2$<br>(chlorimide)<br>[3400-09-7]<br>JSYGRUBHOCKMGQ-UHFFFAOYSA-N | $2.9\times10^{-1}$<br>$2.9\times10^{-1}$<br>$2.9\times10^{-1}$<br>$2.9\times10^{-1}$<br>$2.8\times10^{-1}$ | 4200<br>4200<br>4200<br>4200<br>4200 | Burkholder et al. (2019)<br>Burkholder et al. (2015)<br>Sander et al. (2011)<br>Sander et al. (2006)<br>Holzwarth et al. (1984) | L<br>L<br>L<br>L<br>M | |
| nitrogen trichloride<br>NCl$_3$<br>[10025-85-1]<br>QEHKBHWEUPXBCW-UHFFFAOYSA-N | $9.9\times10^{-4}$<br>$9.9\times10^{-4}$<br>$9.9\times10^{-4}$<br>$9.9\times10^{-4}$<br>$9.9\times10^{-4}$ | 4100<br>4100<br>4100<br>4100<br>4100 | Burkholder et al. (2019)<br>Burkholder et al. (2015)<br>Sander et al. (2011)<br>Sander et al. (2006)<br>Holzwarth et al. (1984) | L<br>L<br>L<br>L<br>M | |



### A1.6 Bromine (Br)

Table A1.6: Bromine (Br)

| Substance Formula (Trivial Name) [CAS Registry Number] InChIKey | $H_s^{cp}$ (at $T^{\ominus}$) $\left[\dfrac{\mathrm{mol}}{\mathrm{m^3\,Pa}}\right]$ | $\dfrac{\mathrm{d\ln} H_s^{cp}}{\mathrm{d}(1/T)}$ [K] | Reference | Type | Note |
|---|---|---|---|---|---|
| bromine (molecular) | $7.2\times10^{-3}$ | 4400 | Burkholder et al. (2019) | L | |
| Br$_2$ | $7.2\times10^{-3}$ | 4400 | Burkholder et al. (2015) | L | |
| [7726-95-6] | $7.2\times10^{-3}$ | 4400 | Sander et al. (2011) | L | |
| GDTBXPJZTBHREO-UHFFFAOYSA-N | $7.2\times10^{-3}$ | 4400 | Sander et al. (2006) | L | |
| | $1.8\times10^{-2}$ | 3600 | Dubik et al. (1987) | M | 142 |
| | $6.8\times10^{-3}$ | | Hill et al. (1968) | M | |
| | $9.6\times10^{-3}$ | | Jenkins and King (1965) | M | 12 |
| | $7.0\times10^{-3}$ | 4100 | Kelley and Tartar (1956) | M | |
| | $1.4\times10^{-2}$ | | Jones (1911) | M | 80 |
| | $7.8\times10^{-3}$ | 3800 | Winkler (1906) | M | |
| | $7.9\times10^{-3}$ | 3600 | Winkler (1899) | M | |
| | $8.3\times10^{-3}$ | 4100 | Fogg and Sangster (2003) | V | |
| | $7.9\times10^{-3}$ | 3900 | Jenkins and King (1965) | R | |
| | $7.2\times10^{-3}$ | 4000 | Wagman et al. (1982) | T | |
| | $9.2\times10^{-3}$ | | Giona et al. (1969) | X | 143, 12 |
| | $7.6\times10^{-3}$ | | Bartlett and Margerum (1999) | ? | 21, 117 |
| | $7.5\times10^{-3}$ | 3900 | Dean and Lange (1999) | ? | 144, 23 |
| bromine atom | $1.2\times10^{-2}$ | | Mozurkewich (1986) | T | 119 |
| Br | $3.4\times10^{-4}$ | 1800 | Berdnikov and Bazhin (1970) | T | 47 |
| [10097-32-2] | | | | | |
| WKBOTKDWSSQWDR-UHFFFAOYSA-N | | | | | |
| hydrogen bromide | | | Carslaw et al. (1995) | T | 145, 1 |
| HBr | | | Brimblecombe and Clegg (1989) | T | 146 |
| [10035-10-6] | | | Wagman et al. (1982) | T | 147 |
| CPELXLSAUQHCOX-UHFFFAOYSA-N | $6.8\times10^{-2}$ | | Hayer et al. (2022) | Q | 20 |
| | | | Chameides and Stelson (1992) | ? | 148 |
| | $2.4\times10^{-1}$ | 250 | Dean and Lange (1999) | ? | 149, 23 |
| | | | Brimblecombe and Clegg (1988) | W | 104 |
| hypobromous acid | $>1.3\times10^{1}$ | | Burkholder et al. (2019) | L | |
| HOBr | $>1.3\times10^{1}$ | | Burkholder et al. (2015) | L | |
| [13517-11-8] | $>1.9\times10^{1}$ | | Blatchley et al. (1992) | M | 12 |
| CUILPNURFADTPE-UHFFFAOYSA-N | $1.9\times10^{1}$ | | McCoy et al. (1990) | M | 12 |
| | $1.8\times10^{-2}$ | 4000 | Mozurkewich (1995) | T | 150 |
| | $6.0\times10^{1}$ | | Frenzel et al. (1998) | E | |
| | $9.1\times10^{-1}$ | | Vogt et al. (1996) | E | |
| | | | Sander et al. (2011) | W | 151 |
| | | | Sander et al. (2006) | W | 151 |
| | | | Fickert (1998) | W | 152 |
| nitryl bromide | $3.0\times10^{-3}$ | | Frenzel et al. (1998) | E | |
| BrNO$_2$ | | | | | |
| [13536-70-4] | | | | | |
| SEYAFCXQVVHRPY-UHFFFAOYSA-M | | | | | |





Table A1.6: Bromine (Br) (...continued)

| Substance<br>Formula<br>(Trivial Name)<br>[CAS Registry Number]<br>InChIKey | $H_s^{cp}$<br>(at $T^\ominus$)<br>$\left[\dfrac{\text{mol}}{\text{m}^3\,\text{Pa}}\right]$ | $\dfrac{\text{d}\ln H_s^{cp}}{\text{d}(1/T)}$<br><br>[K] | Reference | Type | Note |
|---|---|---|---|---|---|
| bromine nitrate<br>BrNO$_3$<br>[40423-14-1]<br>RRTWEEAEXPZMPY-UHFFFAOYSA-N | $\infty$ | | Sander and Crutzen (1996) | E | 98 |
| bromine chloride<br>BrCl<br>[13863-41-7]<br>CODNYICXDISAEA-UHFFFAOYSA-N | $9.7\times10^{-3}$ | 5600 | Burkholder et al. (2019) | L | |
| | $9.7\times10^{-3}$ | 5600 | Burkholder et al. (2015) | L | |
| | $9.7\times10^{-3}$ | 5600 | Sander et al. (2011) | L | |
| | $9.7\times10^{-3}$ | 5600 | Sander et al. (2006) | L | |
| | $<6.2\times10^{-2}$ | | Katrib et al. (2001) | M | 153 |
| | $1.5\times10^{-2}$ | | Disselkamp et al. (1999) | M | 154 |
| | $9.3\times10^{-3}$ | 5600 | Bartlett and Margerum (1999) | M | |
| | $4.2\times10^{-2}$ | 4000 | Dubik et al. (1987) | M | 142 |
| | $1.1\times10^{-2}$ | | this work | T | 155 |
| | $6.9\times10^{-4}$ | 4000 | Ordóñez et al. (2012) | E | |
| | $5.8\times10^{-3}$ | | Frenzel et al. (1998) | E | |



### A1.7 Iodine (I)

Table A1.7: Iodine (I)

| Substance Formula (Trivial Name) [CAS Registry Number] InChIKey | $H_s^{cp}$ (at $T^\ominus$) $\left[\dfrac{\mathrm{mol}}{\mathrm{m^3\,Pa}}\right]$ | $\dfrac{\mathrm{d}\ln H_s^{cp}}{\mathrm{d}(1/T)}$ [K] | Reference | Type | Note |
|---|---|---|---|---|---|
| iodine (molecular) | $2.8\times10^{-2}$ | 4300 | Eguchi et al. (1973) | M | |
| I$_2$ | $2.8\times10^{-2}$ | 3900 | Fogg and Sangster (2003) | V | |
| [7553-56-2] | $3.0\times10^{-2}$ | 4400 | Palmer et al. (1985) | R | 1 |
| PNDPGZBMCMUPRI-UHFFFAOYSA-N | $3.1\times10^{-2}$ | 4600 | Berdnikov and Bazhin (1970) | R | |
| | $3.2\times10^{-2}$ | 4800 | Wagman et al. (1982) | T | |
| | $1.1\times10^{-2}$ | | Thompson and Zafiriou (1983) | C | 156 |
| iodine atom | $7.9\times10^{-4}$ | | Mozurkewich (1986) | T | 157 |
| I | $6.2\times10^{-5}$ | 2300 | Berdnikov and Bazhin (1970) | T | 47 |
| [14362-44-8] | | | | | |
| ZCYVEMRRCGMTRW-UHFFFAOYSA-N | | | | | |
| hydrogen iodide | | | Brimblecombe and Clegg (1989) | T | 158 |
| HI | | | Wagman et al. (1982) | T | 159 |
| [10034-85-2] | $2.5\times10^{-1}$ | 9800 | Ordóñez et al. (2012) | E | |
| XMBWDFGMSWQBCA-UHFFFAOYSA-N | $\infty$ | | Vogt et al. (1999) | E | 98 |
| | | | Brimblecombe and Clegg (1988) | W | 104 |
| iodine monoxide | 4.4 | | Saiz-Lopez et al. (2014) | ? | 160 |
| IO | | | | | |
| [14696-98-1] | | | | | |
| AFSVSXMRDKPOEW-UHFFFAOYSA-N | | | | | |
| iodine dioxide | $9.9\times10^1$ | | Saiz-Lopez et al. (2014) | ? | 160 |
| OIO | | | | | |
| [13494-92-3] | | | | | |
| WXDJHDMIIZKXSK-UHFFFAOYSA-N | | | | | |
| diiodine dioxide | $\infty$ | | Badia et al. (2019) | E | 161, 162 |
| I$_2$O$_2$ | $\infty$ | | Vogt et al. (1999) | E | 161, 98 |
| [215239-62-6] | $9.9\times10^1$ | | Saiz-Lopez et al. (2014) | ? | 161, 160 |
| IELAHHPSAVYAOC-UHFFFAOYSA-N | | | | | |
| diiodine trioxide | $\infty$ | | Badia et al. (2019) | E | 161, 162 |
| I$_2$O$_3$ | $9.9\times10^1$ | | Saiz-Lopez et al. (2014) | ? | 161, 160 |
| [11085-17-9] | | | | | |
| NMNCVPKLBXOKQA-UHFFFAOYSA-N | | | | | |
| diiodine tetroxide | $\infty$ | | Badia et al. (2019) | E | 161, 162 |
| I$_2$O$_4$ | $9.9\times10^1$ | | Saiz-Lopez et al. (2014) | ? | 161, 160 |
| [1024652-24-1] | | | | | |
| XHTWXUOEQMOFEJ-UHFFFAOYSA-N | | | | | |
| hypoiodous acid | $>4.1$ | | Palmer et al. (1985) | C | |
| HOI | | | Thompson and Zafiriou (1983) | E | 163 |
| [14332-21-9] | | | | | |
| GEOVEUCEIQCBKH-UHFFFAOYSA-N | | | | | |





Table A1.7: Iodine (I) (...continued)

| Substance Formula (Trivial Name) [CAS Registry Number] InChIKey | $H_s^{cp}$ (at $T^\ominus$) $\left[\dfrac{\text{mol}}{\text{m}^3\,\text{Pa}}\right]$ | $\dfrac{\text{d}\ln H_s^{cp}}{\text{d}(1/T)}$ [K] | Reference | Type | Note |
|---|---|---|---|---|---|
| iodine nitrite INO$_2$ [15465-40-4] PSZTWRLUACEPOW-UHFFFAOYSA-N | $3.0\times10^{-3}$ $\infty$ | | Badia et al. (2019) Vogt et al. (1999) | E E | 164 98 |
| iodine nitrate INO$_3$ [14696-81-2] CCJHDZZUWZIVJF-UHFFFAOYSA-N | $\infty$ | | Vogt et al. (1999) | E | 98 |
| iodine chloride ICl [7790-99-0] QZRGKCOWNLSUDK-UHFFFAOYSA-N | 1.1 | | Wagman et al. (1982) | T | |
| iodine bromide IBr [7789-33-5] CBEQRNSPHCCXSH-UHFFFAOYSA-N | $2.4\times10^{-1}$ | | Wagman et al. (1982) | T | |



## A1.8 Sulfur (S)

Table A1.8: Sulfur (S)

| Substance Formula (Trivial Name) [CAS Registry Number] InChIKey | $H_s^{cp}$ (at $T^\ominus$) $\left[\dfrac{\mathrm{mol}}{\mathrm{m^3\,Pa}}\right]$ | $\dfrac{\mathrm{d}\ln H_s^{cp}}{\mathrm{d}(1/T)}$ [K] | Reference | Type | Note |
|---|---|---|---|---|---|
| sulfur S [7704-34-9] NINIDFKCEFEMDL-UHFFFAOYSA-N | $2.0\times10^1$ | | Maniere et al. (2011) | ? | 12, 165 |
| hydrogen sulfide H$_2$S [7783-06-4] RWSOTUBLDIXVET-UHFFFAOYSA-N | $1.0\times10^{-3}$ | 2100 | Burkholder et al. (2019) | L | 1 |
| | $1.0\times10^{-3}$ | 2100 | Burkholder et al. (2015) | L | 1 |
| | $1.0\times10^{-3}$ | 2100 | Sander et al. (2011) | L | 1 |
| | $1.0\times10^{-3}$ | 2100 | Sander et al. (2006) | L | 1 |
| | $1.0\times10^{-3}$ | 2000 | Fernández-Prini et al. (2003) | L | 3 |
| | $1.0\times10^{-3}$ | 2200 | Carroll and Mather (1989) | L | |
| | $1.0\times10^{-3}$ | 2000 | Fogg and Young (1988) | L | 1, 166 |
| | $1.0\times10^{-3}$ | 2000 | Yoo et al. (1986) | L | 1 |
| | $1.0\times10^{-3}$ | 2100 | Edwards et al. (1978) | L | 1 |
| | $1.0\times10^{-3}$ | 2100 | Wilhelm et al. (1977) | L | |
| | $9.1\times10^{-4}$ | 1700 | Rinker and Sandall (2000) | M | |
| | $9.2\times10^{-4}$ | 1600 | Munder et al. (2000) | M | |
| | $8.6\times10^{-4}$ | 2100 | De Bruyn et al. (1995b) | M | |
| | $1.1\times10^{-3}$ | 2300 | Suleimenov and Krupp (1994) | M | 1 |
| | $1.2\times10^{-3}$ | 1700 | Tsuji et al. (1990) | M | 62 |
| | $9.4\times10^{-4}$ | 2300 | Barrett et al. (1988) | M | |
| | $1.0\times10^{-3}$ | 2100 | Clarke and Glew (1971) | M | 167 |
| | $1.0\times10^{-3}$ | 2300 | Winkler (1907) | M | |
| | $1.0\times10^{-3}$ | 2100 | Winkler (1906) | M | |
| | $1.1\times10^{-3}$ | 2000 | Schoenfeld (1855) | M | 168 |
| | $9.6\times10^{-4}$ | 2000 | Iliuta and Larachi (2007) | R | 1 |
| | $1.0\times10^{-3}$ | | Hine and Weimar (1965) | R | |
| | $1.0\times10^{-3}$ | 2300 | Edwards et al. (1975) | T | 1 |
| | $7.0\times10^{-4}$ | | Hayer et al. (2022) | Q | 20 |
| | $1.0\times10^{-3}$ | 2000 | Yaws et al. (1999) | ? | 21 |
| | $1.0\times10^{-3}$ | 2100 | Dean and Lange (1999) | ? | 169, 23 |
| | | | Chapoy et al. (2005) | W | 170, 1 |
| deuterium sulfide D$_2$S [13536-94-2] RWSOTUBLDIXVET-ZSJDYOACSA-N | $9.9\times10^{-4}$ | 2100 | Clarke and Glew (1971) | M | 171, 172 |
| sulfur dioxide SO$_2$ [7446-09-5] RAHZWNYVWXNFOC-UHFFFAOYSA-N | $1.3\times10^{-2}$ | 2900 | Burkholder et al. (2019) | L | 1 |
| | $1.3\times10^{-2}$ | 2900 | Burkholder et al. (2015) | L | 1 |
| | $1.3\times10^{-2}$ | 2900 | Sander et al. (2011) | L | 1 |
| | $1.3\times10^{-2}$ | 2900 | Sander et al. (2006) | L | 1 |
| | $1.2\times10^{-2}$ | 3100 | Yoo et al. (1986) | L | 1 |
| | $1.3\times10^{-2}$ | 2900 | Young (1983) | L | 1 |
| | $1.2\times10^{-2}$ | 3200 | Maahs (1982) | L | |
| | $1.2\times10^{-2}$ | 3000 | Edwards et al. (1978) | L | 1 |



Table A1.8: Sulfur (S) (. . . continued)

| Substance<br>Formula<br>(Trivial Name)<br>[CAS Registry Number]<br>InChIKey | $H_s^{cp}$<br>(at $T^{\ominus}$)<br>$\left[\dfrac{\text{mol}}{\text{m}^3\,\text{Pa}}\right]$ | $\dfrac{\text{d}\ln H_s^{cp}}{\text{d}(1/T)}$<br><br>[K] | Reference | Type | Note |
|---|---|---|---|---|---|
| | $1.4\times10^{-2}$ | 2800 | Wilhelm et al. (1977) | L | |
| | $4.0\times10^{-1}$ | | St-Pierre et al. (2014) | M | 173 |
| | $1.2\times10^{-2}$ | 3100 | Johnstone and Leppla (1934) | M | |
| | $1.4\times10^{-2}$ | 3400 | Schoenfeld (1855) | M | 174 |
| | $1.1\times10^{-2}$ | 1200 | Terraglio and Manganelli (1967) | V | |
| | $1.2\times10^{-2}$ | 3100 | Chameides (1984) | T | |
| | $1.2\times10^{-2}$ | 3100 | Edwards et al. (1975) | T | 1 |
| | $1.2\times10^{-2}$ | | Rodríguez-Sevilla et al. (2001) | X | 175 |
| | $1.2\times10^{-2}$ | 3100 | Pandis and Seinfeld (1989) | C | |
| | $1.2\times10^{-2}$ | 3300 | Beilke and Gravenhorst (1978) | C | |
| | $2.9\times10^{-2}$ | | Hayer et al. (2022) | Q | 20 |
| | $1.3\times10^{-2}$ | 2900 | Yaws et al. (1999) | ? | 21 |
| | $1.5\times10^{-2}$ | 3100 | Dean and Lange (1999) | ? | 176, 23 |
| | $1.2\times10^{-2}$ | 3100 | Seinfeld (1986) | ? | 21 |
| | $1.2\times10^{-2}$ | 3100 | Hoffmann and Jacob (1984) | ? | 21 |
| sulfur trioxide<br>SO$_3$<br>[7446-11-9]<br>AKEJUJNQAAGONA-UHFFFAOYSA-N | $5.1\times10^{-8}$<br>$\infty$ | | Hayer et al. (2022)<br>Sander and Crutzen (1996) | Q<br>E | 20<br>98 |
| sulfuric acid<br>H$_2$SO$_4$<br>[7664-93-9]<br>QAOWNCQODCNURD-UHFFFAOYSA-N | <br><br><br><br>$1.3\times10^{13}$<br>$2.9\times10^{7}$ | <br><br><br><br>20000<br>10000 | Marti et al. (1997)<br>Ayers et al. (1980)<br>Gmitro and Vermeulen (1964)<br>Clegg et al. (1998)<br>Hoffmann and Calvert (1985)<br>Ayers (1983) | M<br>M<br>M<br>V<br>T<br>T | 177<br>178<br>179<br>180<br><br> |
| sulfur hexafluoride<br>SF$_6$<br>[2551-62-4]<br>SFZCNBIFKDRMGX-UHFFFAOYSA-N | $2.4\times10^{-6}$ | 3100 | Warneck and Williams (2012) | L | |
| | $2.5\times10^{-6}$ | 2100 | Fernández-Prini et al. (2003) | L | 3 |
| | $2.4\times10^{-6}$ | 2400 | Wilhelm et al. (1977) | L | |
| | $2.3\times10^{-6}$ | 2700 | Bullister et al. (2002) | M | 181 |
| | $1.4\times10^{-6}$ | | Guitart et al. (1989) | M | 14 |
| | $2.4\times10^{-6}$ | | Park et al. (1982) | M | |
| | $2.5\times10^{-6}$ | 260 | Cosgrove and Walkley (1981) | M | 42, 11 |
| | $1.7\times10^{-6}$ | | Longo et al. (1970) | M | 14 |
| | $1.7\times10^{-6}$ | | Power and Stegall (1970) | M | 14 |
| | | | Shoor et al. (1969) | M | 182 |
| | $2.4\times10^{-6}$ | 2400 | Ashton et al. (1968) | M | 183 |
| | $2.2\times10^{-6}$ | 3500 | Friedman (1954) | M | |
| | $2.2\times10^{-6}$ | | Giardino et al. (1988) | V | |
| | $2.6\times10^{-6}$ | | Hayer et al. (2022) | Q | 20 |
| | $2.2\times10^{-5}$ | | Keshavarz et al. (2022) | Q | |
| | $2.6\times10^{-5}$ | | Duchowicz et al. (2020) | Q | 184 |
| | | 3200 | Kühne et al. (2005) | Q | |
| | $2.2\times10^{-6}$ | | Duchowicz et al. (2020) | ? | 185, 21 |
| | | 2800 | Kühne et al. (2005) | ? | |
| | $2.4\times10^{-6}$ | 2400 | Yaws et al. (1999) | ? | 21 |
| | $1.9\times10^{-6}$ | | Abraham and Weathersby (1994) | ? | 21 |





Table A1.8: Sulfur (S) (...continued)

| Substance Formula (Trivial Name) [CAS Registry Number] InChIKey | $H_s^{cp}$ (at $T^\ominus$) $\left[\dfrac{\text{mol}}{\text{m}^3\,\text{Pa}}\right]$ | $\dfrac{\text{d}\ln H_s^{cp}}{\text{d}(1/T)}$ [K] | Reference | Type | Note |
|---|---|---|---|---|---|
| sulfuryl fluoride | $8.9\times10^{-5}$ | 3100 | Cady and Misra (1974) | M | |
| SO$_2$F$_2$ | $7.3\times10^{-5}$ | | Duchowicz et al. (2020) | V | 186 |
| [2699-79-8] | 3.2 | | Duchowicz et al. (2020) | Q | |
| OBTWBSRJZRCYQV-UHFFFAOYSA-N | $6.3\times10^{-6}$ | | Maniere et al. (2011) | ? | 12, 165 |



### A1.9    Rare gases (He, Ne, Ar, Kr, Xe, Rn)

Table A1.9: Rare gases (He, Ne, Ar, Kr, Xe, Rn)

| Substance / Formula / (Trivial Name) / [CAS Registry Number] / InChIKey | $H_s^{cp}$ (at $T^{\ominus}$) $\left[\dfrac{\mathrm{mol}}{\mathrm{m^3\,Pa}}\right]$ | $\dfrac{\mathrm{d}\ln H_s^{cp}}{\mathrm{d}(1/T)}$ [K] | Reference | Type | Note |
|---|---|---|---|---|---|
| helium | $3.9\times10^{-6}$ | 15 | Fernández-Prini et al. (2003) | L | 3 |
| He | $3.8\times10^{-6}$ | 83 | Abraham and Matteoli (1988) | L | |
| [7440-59-7] | $3.8\times10^{-6}$ | 83 | Clever (1979a) | L | 1 |
| SWQJXJOGLNCZEY-UHFFFAOYSA-N | $3.8\times10^{-6}$ | 92 | Wilhelm et al. (1977) | L | |
| | $3.7\times10^{-6}$ | 320 | Himmelblau (1960) | L | 1 |
| | $3.9\times10^{-6}$ | 69 | Krause and Benson (1989) | M | |
| | | | Shoor et al. (1969) | M | 187 |
| | $3.7\times10^{-6}$ | 120 | Morrison and Johnstone (1954) | M | 188 |
| | $3.8\times10^{-6}$ | | Friedman (1954) | M | |
| | $3.8\times10^{-6}$ | 210 | Lannung (1930) | M | 189 |
| | $3.7\times10^{-6}$ | 380 | Cady et al. (1922) | M | |
| | $6.3\times10^{-6}$ | -700 | von Antropoff (1910) | M | 42 |
| | $3.7\times10^{-6}$ | 220 | Wauchope and Haque (1972) | V | |
| | $5.3\times10^{-6}$ | | Pierotti (1965) | T | |
| | $4.5\times10^{-6}$ | | Hayer et al. (2022) | Q | 20 |
| | $3.3\times10^{-6}$ | 4 | Linnemann et al. (2020) | Q | 42 |
| | $3.3\times10^{-6}$ | 71 | Linnemann et al. (2020) | Q | 42, 190 |
| | $3.9\times10^{-6}$ | | Warr et al. (2015) | Q | 12 |
| | $3.8\times10^{-6}$ | 83 | Yaws et al. (1999) | ? | 21 |
| | $3.9\times10^{-6}$ | | Abraham and Weathersby (1994) | ? | 21 |
| | $3.7\times10^{-6}$ | 200 | Dean and Lange (1999) | ? | 191, 23 |
| | $3.8\times10^{-6}$ | | Abraham et al. (1990) | ? | |
| neon | $4.5\times10^{-6}$ | 430 | Fernández-Prini et al. (2003) | L | 3 |
| Ne | $4.4\times10^{-6}$ | 470 | Abraham and Matteoli (1988) | L | |
| [7440-01-9] | $4.5\times10^{-6}$ | 470 | Clever (1979a) | L | 1 |
| GKAOGPIIYCISHV-UHFFFAOYSA-N | $4.4\times10^{-6}$ | 450 | Wilhelm et al. (1977) | L | |
| | $4.5\times10^{-6}$ | 440 | Krause and Benson (1989) | M | |
| | $4.4\times10^{-6}$ | 510 | Crovetto et al. (1982) | M | |
| | $4.3\times10^{-6}$ | | Power and Stegall (1970) | M | 14 |
| | $4.5\times10^{-6}$ | 460 | Morrison and Johnstone (1954) | M | 192 |
| | $4.6\times10^{-6}$ | 37 | Lannung (1930) | M | 193 |
| | $6.6\times10^{-6}$ | -990 | von Antropoff (1910) | M | |
| | $4.5\times10^{-6}$ | 510 | Wauchope and Haque (1972) | V | |
| | $8.8\times10^{-6}$ | | Pierotti (1965) | T | |
| | $4.5\times10^{-6}$ | | Hayer et al. (2022) | Q | 20 |
| | $3.4\times10^{-6}$ | 250 | Linnemann et al. (2020) | Q | 33 |
| | $4.7\times10^{-6}$ | 470 | Linnemann et al. (2020) | Q | 33, 190 |
| | $3.6\times10^{-6}$ | | Warr et al. (2015) | Q | 12 |
| | $4.5\times10^{-6}$ | 470 | Yaws et al. (1999) | ? | 21 |
| | $4.4\times10^{-6}$ | | Abraham and Weathersby (1994) | ? | 21 |
| | $4.5\times10^{-6}$ | 550 | Dean and Lange (1999) | ? | 194, 23 |
| | $4.4\times10^{-6}$ | | Abraham et al. (1990) | ? | |



Table A1.9: Rare gases (He, Ne, Ar, Kr, Xe, Rn) (. . . continued)

| Substance Formula (Trivial Name) [CAS Registry Number] InChIKey | $H_s^{cp}$ (at $T^{\ominus}$) $\left[\dfrac{\text{mol}}{\text{m}^3\,\text{Pa}}\right]$ | $\dfrac{\text{d}\ln H_s^{cp}}{\text{d}(1/T)}$ [K] | Reference | Type | Note |
|---|---|---|---|---|---|
| argon | $1.4\times10^{-5}$ | 1700 | Warneck and Williams (2012) | L | |
| Ar | $1.4\times10^{-5}$ | 1400 | Fernández-Prini et al. (2003) | L | 3 |
| [7440-37-1] | $1.4\times10^{-5}$ | 1500 | Abraham and Matteoli (1988) | L | |
| XKRFYHLGVUSROY-UHFFFAOYSA-N | $1.4\times10^{-5}$ | 1500 | Clever (1980) | L | 1 |
| | $1.4\times10^{-5}$ | 1500 | Wilhelm et al. (1977) | L | |
| | $1.4\times10^{-5}$ | 1400 | Rettich et al. (1992) | M | 195 |
| | $1.4\times10^{-5}$ | 1400 | Krause and Benson (1989) | M | |
| | $1.4\times10^{-5}$ | | Park et al. (1982) | M | |
| | $1.4\times10^{-5}$ | 1500 | Crovetto et al. (1982) | M | |
| | $1.4\times10^{-5}$ | 1200 | Cosgrove and Walkley (1981) | M | 11 |
| | $1.4\times10^{-5}$ | 1300 | Potter II and Clynne (1978) | M | |
| | $1.4\times10^{-5}$ | 1500 | Murray and Riley (1970) | M | 196 |
| | $1.4\times10^{-5}$ | 1600 | Shoor et al. (1969) | M | 197 |
| | $1.4\times10^{-5}$ | 1500 | Ashton et al. (1968) | M | 198 |
| | $1.3\times10^{-5}$ | 1500 | Morrison and Johnstone (1954) | M | 199 |
| | $1.4\times10^{-5}$ | 1800 | Friedman (1954) | M | |
| | $1.4\times10^{-5}$ | 1400 | Lannung (1930) | M | 200 |
| | $1.6\times10^{-5}$ | 1300 | von Antropoff (1910) | M | |
| | $1.5\times10^{-5}$ | 1400 | Winkler (1906) | M | |
| | $1.4\times10^{-5}$ | 1400 | Wauchope and Haque (1972) | V | |
| | $1.4\times10^{-5}$ | 1400 | Wauchope and Haque (1972) | V | |
| | $1.8\times10^{-5}$ | | Pierotti (1965) | T | |
| | $9.5\times10^{-6}$ | | Hayer et al. (2022) | Q | 20 |
| | $7.8\times10^{-6}$ | 1200 | Linnemann et al. (2020) | Q | 190 |
| | $1.1\times10^{-5}$ | 1100 | Linnemann et al. (2020) | Q | 201 |
| | $1.2\times10^{-5}$ | | Warr et al. (2015) | Q | 12 |
| | $1.4\times10^{-5}$ | 1500 | Yaws et al. (1999) | ? | 21 |
| | $1.2\times10^{-5}$ | | Abraham and Weathersby (1994) | ? | 21 |
| | $1.4\times10^{-5}$ | 1500 | Dean and Lange (1999) | ? | 202, 23 |
| | $1.4\times10^{-5}$ | | Abraham et al. (1990) | ? | |
| krypton | $2.5\times10^{-5}$ | 1700 | Fernández-Prini et al. (2003) | L | 3 |
| Kr | $2.5\times10^{-5}$ | 1900 | Abraham and Matteoli (1988) | L | |
| [7439-90-9] | $2.5\times10^{-5}$ | 1900 | Clever (1979b) | L | 1 |
| DNNSSWSSYDEUBZ-UHFFFAOYSA-N | $2.5\times10^{-5}$ | 1900 | Wilhelm et al. (1977) | L | |
| | $2.0\times10^{-5}$ | | Steward et al. (1973) | L | 14 |
| | $2.5\times10^{-5}$ | 2000 | Allott et al. (1973) | L | |
| | $2.5\times10^{-5}$ | 1800 | Krause and Benson (1989) | M | |
| | $2.5\times10^{-5}$ | 1900 | Crovetto et al. (1982) | M | |
| | $2.6\times10^{-5}$ | 1800 | Cosgrove and Walkley (1981) | M | 11 |
| | $2.4\times10^{-5}$ | 1700 | Morrison and Johnstone (1954) | M | 203 |
| | $3.4\times10^{-5}$ | 1400 | von Antropoff (1910) | M | 204 |
| | $2.8\times10^{-5}$ | 1900 | von Antropoff (1910) | M | 204 |
| | $2.4\times10^{-5}$ | 1800 | Wauchope and Haque (1972) | V | |
| | $4.4\times10^{-5}$ | | Pierotti (1965) | T | |
| | $2.6\times10^{-5}$ | | Hayer et al. (2022) | Q | 20 |
| | $1.5\times10^{-5}$ | 1600 | Linnemann et al. (2020) | Q | 190 |
| | $1.6\times10^{-5}$ | 1400 | Linnemann et al. (2020) | Q | 201 |



Table A1.9: Rare gases (He, Ne, Ar, Kr, Xe, Rn) (...continued)

| Substance Formula (Trivial Name) [CAS Registry Number] InChIKey | $H_s^{cp}$ (at $T^{\ominus}$) $\left[\dfrac{\mathrm{mol}}{\mathrm{m}^3\,\mathrm{Pa}}\right]$ | $\dfrac{\mathrm{d}\ln H_s^{cp}}{\mathrm{d}(1/T)}$ [K] | Reference | Type | Note |
|---|---|---|---|---|---|
| | $2.5\times10^{-5}$ | | Warr et al. (2015) | Q | 12 |
| | $2.5\times10^{-5}$ | 1900 | Yaws et al. (1999) | ? | 21 |
| | $2.0\times10^{-5}$ | | Abraham and Weathersby (1994) | ? | 21 |
| | $2.5\times10^{-5}$ | 1800 | Dean and Lange (1999) | ? | 205, 23 |
| | $2.5\times10^{-5}$ | | Abraham et al. (1990) | ? | |
| xenon Xe [7440-63-3] FHNFHKCVQCLJFQ-UHFFFAOYSA-N | $4.4\times10^{-5}$ | 2200 | Fernández-Prini et al. (2003) | L | 3 |
| | $4.3\times10^{-5}$ | 2300 | Abraham and Matteoli (1988) | L | |
| | $4.3\times10^{-5}$ | 2300 | Clever (1979b) | L | 1 |
| | $4.2\times10^{-5}$ | 2200 | Wilhelm et al. (1977) | L | |
| | $3.3\times10^{-5}$ | | Steward et al. (1973) | L | 14 |
| | $4.5\times10^{-5}$ | 2400 | Allott et al. (1973) | L | |
| | $4.0\times10^{-5}$ | 2400 | Himmelblau (1960) | L | 1, 206 |
| | $4.3\times10^{-5}$ | 2300 | Krause and Benson (1989) | M | |
| | $4.2\times10^{-5}$ | 2400 | Crovetto et al. (1982) | M | |
| | $4.2\times10^{-5}$ | 2200 | Morrison and Johnstone (1954) | M | 207 |
| | $4.4\times10^{-5}$ | 2500 | von Antropoff (1910) | M | |
| | $4.2\times10^{-5}$ | 2200 | Wauchope and Haque (1972) | V | |
| | $5.5\times10^{-5}$ | | Pierotti (1965) | T | |
| | $2.5\times10^{-5}$ | | Hayer et al. (2022) | Q | 20 |
| | $7.0\times10^{-5}$ | 2300 | Linnemann et al. (2020) | Q | 190 |
| | $2.9\times10^{-5}$ | 1800 | Linnemann et al. (2020) | Q | 201 |
| | $8.2\times10^{-5}$ | | Warr et al. (2015) | Q | 12 |
| | $4.3\times10^{-5}$ | 2300 | Yaws et al. (1999) | ? | 21 |
| | $3.4\times10^{-5}$ | | Abraham and Weathersby (1994) | ? | 21 |
| | $4.9\times10^{-5}$ | 2200 | Dean and Lange (1999) | ? | 208, 23 |
| | $4.3\times10^{-5}$ | | Abraham et al. (1990) | ? | |
| xenon-133 $^{133}$Xe [14932-42-4] FHNFHKCVQCLJFQ-NJFSPNSNSA-N | $3.2\times10^{-5}$ | | Ercan (1979) | M | 14 |
| radon Rn [10043-92-2] SYUHGPGVQRZVTB-UHFFFAOYSA-N | $9.1\times10^{-5}$ | 2900 | Abraham and Matteoli (1988) | L | |
| | $9.1\times10^{-5}$ | 2600 | Clever (1979b) | L | 1 |
| | $9.2\times10^{-5}$ | 2600 | Wilhelm et al. (1977) | L | |
| | $9.4\times10^{-5}$ | 2600 | Lewis et al. (1987) | M | 209 |
| | $1.0\times10^{-4}$ | 2700 | Ramstedt (1911) | M | |
| | $1.2\times10^{-4}$ | | Pierotti (1965) | T | |
| | $1.0\times10^{-4}$ | | Hayer et al. (2022) | Q | 20 |
| | $7.4\times10^{-5}$ | 2400 | Linnemann et al. (2020) | Q | 210 |
| | $9.1\times10^{-5}$ | 2600 | Yaws et al. (1999) | ? | 21 |
| | $8.3\times10^{-5}$ | 2800 | Dean and Lange (1999) | ? | 211, 23 |
| | $9.1\times10^{-5}$ | | Abraham et al. (1990) | ? | |



### A1.10   Other elements (B, Se, P, As, Hg)

Table A1.10: Other elements (B, Se, P, As, Hg)

| Substance Formula (Trivial Name) [CAS Registry Number] InChIKey | $H_s^{cp}$ (at $T^{\ominus}$) $\left[\dfrac{\mathrm{mol}}{\mathrm{m^3\,Pa}}\right]$ | $\dfrac{\mathrm{d}\ln H_s^{cp}}{\mathrm{d}(1/T)}$ [K] | Reference | Type | Note |
|---|---|---|---|---|---|
| boric acid $H_3BO_3$ [10043-35-3] KGBXLFKZBHKPEV-UHFFFAOYSA-N | $3.8\times10^6$ | | HSDB (2015) | V | |
| selenium hydride $H_2Se$ [7783-07-5] BUGBHKTXTAQXES-UHFFFAOYSA-N | $8.1\times10^{-4}$ $8.3\times10^{-4}$ $8.3\times10^{-4}$ | 1700 1900 1900 | Fogg and Young (1988) Wilhelm et al. (1977) Sisi et al. (1971) | L L M | 1 212 |
| phosphorus trihydride $PH_3$ (phosphine) [7803-51-2] XYFCBTPGUUZFHI-UHFFFAOYSA-N | $8.1\times10^{-5}$ $5.9\times10^{-5}$ $8.1\times10^{-5}$ | 2000 3000 2000 | Wilhelm et al. (1977) Fu et al. (2013) Yaws et al. (1999) | L M ? | 213 21 |
| arsenic hydride $AsH_3$ (arsine) [7784-42-1] RQNWIZPPADIBDY-UHFFFAOYSA-N | $8.8\times10^{-5}$ | 2100 | Wilhelm et al. (1977) | L | |
| mercury Hg [7439-97-6] QSHDDOUJBYECFT-UHFFFAOYSA-N | $1.3\times10^{-3}$ $1.3\times10^{-3}$ $1.2\times10^{-3}$ $1.1\times10^{-3}$ $1.3\times10^{-3}$ $1.3\times10^{-3}$ $8.7\times10^{-4}$ $1.1\times10^{-3}$ $1.2\times10^{-3}$ $1.4\times10^{-3}$ $1.4\times10^{-3}$ $1.3\times10^{-3}$ $1.3\times10^{-3}$ $9.2\times10^{-4}$ | 2600 2600 5400 4800 2600 2500 5700 2300 2700 | Burkholder et al. (2019) Burkholder et al. (2015) Clever (1987) Clever et al. (1985) Andersson et al. (2008) Sanemasa (1975) Mackay and Leinonen (1975) Glew and Hames (1971) Shon et al. (2005) WHO (1990) Abraham et al. (2008) Schroeder and Munthe (1998) Schroeder and Munthe (1998) Petersen et al. (1998) Brimblecombe (1986) | L L L L M M V V C C Q ? ? ? ? | 1 1 1 12 214 12, 21 21 215 80 |
| mercury(II) oxide HgO [21908-53-2] UKWHYYKOEPRTIC-UHFFFAOYSA-N | $3.2\times10^4$ $2.7\times10^{10}$ $1.4\times10^4$ | | Shon et al. (2005) Schroeder and Munthe (1998) Petersen et al. (1998) | ? ? ? | 216 21 215 |
| mercury dihydroxide $Hg(OH)_2$ [12135-13-6] VLKKXDVIWIBHHS-UHFFFAOYSA-L | $1.3\times10^2$ $1.3\times10^2$ | 4200 4200 | WHO (1990) Lindqvist and Rodhe (1985) | C C | |



Table A1.10: Other elements (B, Se, P, As, Hg) (...continued)

| Substance Formula (Trivial Name) [CAS Registry Number] InChIKey | $H_s^{cp}$ (at $T^{\ominus}$) $\left[\dfrac{\text{mol}}{\text{m}^3\,\text{Pa}}\right]$ | $\dfrac{\text{d}\ln H_s^{cp}}{\text{d}(1/T)}$ [K] | Reference | Type | Note |
|---|---|---|---|---|---|
| mercury dichloride | $1.0\times10^3$ | | Severit (1997) | M | 217 |
| HgCl$_2$ | $1.6\times10^4$ | | Abraham et al. (2008) | V | |
| [7487-94-7] | $4.2\times10^4$ | | Abraham et al. (2008) | V | |
| LWJROJCJINYWOX-UHFFFAOYSA-L | $1.3\times10^4$ | 7400 | Kanefke (2008) | R | |
| | $2.4\times10^5$ | | Shon et al. (2005) | C | |
| | $1.4\times10^4$ | 5300 | WHO (1990) | C | |
| | $1.4\times10^4$ | 5300 | Lindqvist and Rodhe (1985) | C | |
| | $4.2\times10^4$ | 7400 | Abraham et al. (2008) | Q | 218 |
| | $2.7\times10^4$ | | Schroeder and Munthe (1998) | ? | 12, 21 |
| | $1.4\times10^4$ | 9500 | Braun and Dransfeld (1989) | ? | 11 |
| | $6.3\times10^2$ | | Iverfeldt and Persson (1985) | ? | 219 |
| mercury dibromide | $1.2\times10^3$ | | Abraham et al. (2008) | V | |
| HgBr$_2$ | $9.6\times10^2$ | 7400 | Kanefke (2008) | C | |
| [7789-47-1] | $4.4\times10^3$ | 7100 | Abraham et al. (2008) | Q | 218 |
| NGYIMTKLQULBOO-UHFFFAOYSA-L | $2.7\times10^4$ | | Hedgecock et al. (2005) | ? | 220 |
| | $5.2\times10^1$ | | Iverfeldt and Persson (1985) | ? | 219 |
| mercury diiodide | $5.7\times10^1$ | | Abraham et al. (2008) | V | |
| HgI$_2$ | $2.0\times10^2$ | 6700 | Abraham et al. (2008) | Q | 218 |
| [7774-29-0] | 1.9 | | Iverfeldt and Persson (1985) | ? | 219 |
| YFDLHELOZYVNJE-UHFFFAOYSA-L | | | | | |



## A2 Hydrocarbons (C, H)

### A2.1 Alkanes

Table A2.1: Alkanes

| Substance Formula (Trivial Name) [CAS Registry Number] InChIKey | $H_s^{cp}$ (at $T^{\ominus}$) $\left[\dfrac{\mathrm{mol}}{\mathrm{m}^3\,\mathrm{Pa}}\right]$ | $\dfrac{\mathrm{d}\ln H_s^{cp}}{\mathrm{d}(1/T)}$ [K] | Reference | Type | Note |
|---|---|---|---|---|---|
| methane | $1.4\times10^{-5}$ | 1600 | Burkholder et al. (2019) | L | 1 |
| $CH_4$ | $1.2\times10^{-5}$ | 1100 | Burkholder et al. (2019) | L | 70 |
| [74-82-8] | $1.4\times10^{-5}$ | 1600 | Burkholder et al. (2015) | L | 1 |
| VNWKTOKETHGBQD-UHFFFAOYSA-N | $1.2\times10^{-5}$ | 1100 | Burkholder et al. (2015) | L | 70 |
| | $1.4\times10^{-5}$ | 1900 | Warneck and Williams (2012) | L | |
| | $1.4\times10^{-5}$ | 1600 | Sander et al. (2011) | L | 1 |
| | $1.4\times10^{-5}$ | 1600 | Sander et al. (2006) | L | 1 |
| | $1.4\times10^{-5}$ | 1500 | Fernández-Prini et al. (2003) | L | 3 |
| | $1.4\times10^{-5}$ | 1600 | Plyasunov and Shock (2000) | L | |
| | $1.4\times10^{-5}$ | 1600 | Abraham and Matteoli (1988) | L | |
| | | | Clever and Young (1987) | L | 221 |
| | $1.5\times10^{-5}$ | | Mackay and Shiu (1981) | L | |
| | $1.4\times10^{-5}$ | 1700 | Wilhelm et al. (1977) | L | |
| | $1.3\times10^{-5}$ | 1500 | Himmelblau (1960) | L | 1 |
| | $1.6\times10^{-5}$ | | Liu et al. (2021) | M | |
| | $1.4\times10^{-5}$ | 1800 | Lutsyk et al. (2005) | M | |
| | $1.2\times10^{-5}$ | 2400 | Lekvam and Bishnoi (1997) | M | |
| | $1.3\times10^{-5}$ | 1400 | Reichl (1995) | M | 222 |
| | $1.4\times10^{-5}$ | 1600 | Scharlin and Battino (1995) | M | 223 |
| | $1.2\times10^{-5}$ | | Guitart et al. (1989) | M | 14 |
| | $1.4\times10^{-5}$ | 1800 | Ben-Naim and Battino (1985) | M | |
| | $1.4\times10^{-5}$ | 1600 | Crovetto et al. (1982) | M | |
| | $1.4\times10^{-5}$ | 1600 | Rettich et al. (1981) | M | |
| | $1.4\times10^{-5}$ | 1600 | Cosgrove and Walkley (1981) | M | 11 |
| | $1.3\times10^{-5}$ | 1700 | Shoor et al. (1969) | M | 224 |
| | $1.5\times10^{-5}$ | | McAuliffe (1966) | M | 225 |
| | $1.4\times10^{-5}$ | 1600 | Wetlaufer et al. (1964) | M | |
| | $1.5\times10^{-5}$ | | McAuliffe (1963) | M | 226 |
| | $1.3\times10^{-5}$ | 1600 | Morrison and Billett (1952) | M | 227 |
| | $1.3\times10^{-5}$ | 1700 | Winkler (1901) | M | 228 |
| | $1.5\times10^{-5}$ | | Duchowicz et al. (2020) | V | 186 |
| | $1.5\times10^{-5}$ | | HSDB (2015) | V | |
| | $1.5\times10^{-5}$ | | Meylan and Howard (1991) | V | |
| | $1.5\times10^{-5}$ | | Hine and Mookerjee (1975) | V | |
| | $1.3\times10^{-5}$ | 1600 | Wauchope and Haque (1972) | V | |
| | $9.2\times10^{-5}$ | | Butler and Ramchandani (1935) | V | |
| | $1.4\times10^{-5}$ | | Hine and Weimar (1965) | R | |
| | $1.4\times10^{-5}$ | | Pierotti (1965) | T | |
| | $9.6\times10^{-6}$ | | Liss and Slater (1974) | C | |
| | $1.3\times10^{-5}$ | | Deno and Berkheimer (1960) | C | |
| | $1.1\times10^{-5}$ | | Hayer et al. (2022) | Q | 20 |
| | $3.4\times10^{-3}$ | | Duchowicz et al. (2020) | Q | |
| | $7.0\times10^{-7}$ | | Gharagheizi et al. (2012) | Q | |



Table A2.1: Alkanes (. . . continued)

| Substance Formula (Trivial Name) [CAS Registry Number] InChIKey | $H_s^{cp}$ (at $T^\ominus$) $\left[\dfrac{\text{mol}}{\text{m}^3\,\text{Pa}}\right]$ | $\dfrac{\mathrm{d}\ln H_s^{cp}}{\mathrm{d}(1/T)}$ [K] | Reference | Type | Note |
|---|---|---|---|---|---|
| | $2.5\times10^{-5}$ | | Hilal et al. (2008) | Q | |
| | | 2300 | Kühne et al. (2005) | Q | |
| | $3.0\times10^{-5}$ | | Yao et al. (2002) | Q | 229 |
| | $3.0\times10^{-5}$ | | English and Carroll (2001) | Q | 230, 231 |
| | $8.6\times10^{-6}$ | | Katritzky et al. (1998) | Q | |
| | $1.6\times10^{-5}$ | | Nirmalakhandan et al. (1997) | Q | |
| | $2.1\times10^{-5}$ | | Suzuki et al. (1992) | Q | 232 |
| | $2.4\times10^{-5}$ | | Meylan and Howard (1991) | Q | |
| | | 1700 | Kühne et al. (2005) | ? | |
| | $1.6\times10^{-5}$ | | Yaws (1999) | ? | 21 |
| | $1.4\times10^{-5}$ | 1600 | Yaws et al. (1999) | ? | 21 |
| | $1.2\times10^{-5}$ | | Abraham and Weathersby (1994) | ? | 21 |
| | $1.3\times10^{-5}$ | 1700 | Dean and Lange (1999) | ? | 233, 23 |
| | $1.5\times10^{-5}$ | | Yaws and Yang (1992) | ? | 21 |
| | $1.4\times10^{-5}$ | | Abraham et al. (1990) | ? | |
| ethane $C_2H_6$ [74-84-0] OTMSDBZUPAUEDD-UHFFFAOYSA-N | $1.9\times10^{-5}$ | 2400 | Burkholder et al. (2019) | L | 1 |
| | $1.9\times10^{-5}$ | 2400 | Burkholder et al. (2015) | L | 1 |
| | $1.9\times10^{-5}$ | 2400 | Sander et al. (2011) | L | 1 |
| | $1.9\times10^{-5}$ | 2400 | Sander et al. (2006) | L | 1 |
| | $1.9\times10^{-5}$ | 2400 | Fernández-Prini et al. (2003) | L | 3 |
| | $1.9\times10^{-5}$ | 2300 | Plyasunov and Shock (2000) | L | |
| | $1.9\times10^{-5}$ | 2300 | Abraham and Matteoli (1988) | L | |
| | $1.9\times10^{-5}$ | 2300 | Hayduk (1982) | L | 1 |
| | $2.0\times10^{-5}$ | | Mackay and Shiu (1981) | L | |
| | $1.8\times10^{-5}$ | 2400 | Wilhelm et al. (1977) | L | |
| | $2.0\times10^{-5}$ | 2300 | Reichl (1995) | M | 234 |
| | $1.3\times10^{-5}$ | | Guitart et al. (1989) | M | 14 |
| | $1.8\times10^{-5}$ | 2700 | Ben-Naim and Battino (1985) | M | |
| | $1.9\times10^{-5}$ | 2300 | Rettich et al. (1981) | M | |
| | $1.8\times10^{-5}$ | 2700 | Cosgrove and Walkley (1981) | M | 11 |
| | $2.0\times10^{-5}$ | | McAuliffe (1966) | M | 225 |
| | $2.0\times10^{-5}$ | 2400 | Wetlaufer et al. (1964) | M | |
| | $2.0\times10^{-5}$ | | McAuliffe (1963) | M | 226 |
| | $1.7\times10^{-5}$ | 2100 | Morrison and Billett (1952) | M | 235 |
| | $1.8\times10^{-5}$ | 2400 | Winkler (1901) | M | 236 |
| | $2.0\times10^{-5}$ | | Duchowicz et al. (2020) | V | 186 |
| | $2.0\times10^{-5}$ | | HSDB (2015) | V | |
| | $2.0\times10^{-5}$ | | Hine and Mookerjee (1975) | V | |
| | $1.7\times10^{-5}$ | 2000 | Wauchope and Haque (1972) | V | |
| | $1.0\times10^{-4}$ | | Butler and Ramchandani (1935) | V | |
| | $4.0\times10^{-5}$ | | Pierotti (1965) | T | |
| | $2.0\times10^{-5}$ | | Yaws (2003) | X | 237 |
| | $1.8\times10^{-5}$ | | Deno and Berkheimer (1960) | C | |
| | $1.8\times10^{-5}$ | | Hayer et al. (2022) | Q | 20 |
| | $1.3\times10^{-3}$ | | Duchowicz et al. (2020) | Q | |
| | $1.1\times10^{-4}$ | | Wang et al. (2017) | Q | 80, 238 |
| | $1.1\times10^{-5}$ | | Wang et al. (2017) | Q | 80, 239 |



Table A2.1: Alkanes (. . . continued)

| Substance<br>Formula<br>(Trivial Name)<br>[CAS Registry Number]<br><small>InChIKey</small> | $H_s^{cp}$<br>(at $T^{\ominus}$)<br>$\left[\dfrac{\mathrm{mol}}{\mathrm{m^3\,Pa}}\right]$ | $\dfrac{\mathrm{d}\ln H_s^{cp}}{\mathrm{d}(1/T)}$<br><br>[K] | Reference | Type | Note |
|---|---|---|---|---|---|
| | $2.0\times10^{-5}$ | | Wang et al. (2017) | Q | 80, 240 |
| | $2.0\times10^{-5}$ | | Li et al. (2014) | Q | 241 |
| | $3.3\times10^{-6}$ | | Gharagheizi et al. (2012) | Q | |
| | $2.0\times10^{-5}$ | | Raventos-Duran et al. (2010) | Q | 242, 243 |
| | $1.2\times10^{-5}$ | | Raventos-Duran et al. (2010) | Q | 244 |
| | $2.0\times10^{-5}$ | | Raventos-Duran et al. (2010) | Q | 245 |
| | $4.1\times10^{-5}$ | | Gharagheizi et al. (2010) | Q | 246 |
| | $2.0\times10^{-5}$ | | Hilal et al. (2008) | Q | |
| | $7.8\times10^{-6}$ | | Modarresi et al. (2007) | Q | 67 |
| | | 2600 | Kühne et al. (2005) | Q | |
| | $4.6\times10^{-6}$ | | Modarresi et al. (2005) | Q | 247 |
| | $2.1\times10^{-5}$ | | Yaffe et al. (2003) | Q | 248, 249 |
| | $2.1\times10^{-5}$ | | Yao et al. (2002) | Q | 229 |
| | $1.8\times10^{-5}$ | | English and Carroll (2001) | Q | 230, 231 |
| | $1.5\times10^{-5}$ | | Katritzky et al. (1998) | Q | |
| | $2.4\times10^{-5}$ | | Suzuki et al. (1992) | Q | 232 |
| | $2.2\times10^{-5}$ | | Nirmalakhandan and Speece (1988) | Q | |
| | $1.1\times10^{-5}$ | | Irmann (1965) | Q | |
| | | 2500 | Kühne et al. (2005) | ? | |
| | $2.0\times10^{-5}$ | | Yaws (1999) | ? | 21 |
| | $1.9\times10^{-5}$ | 2300 | Yaws et al. (1999) | ? | 21 |
| | $1.5\times10^{-5}$ | | Abraham and Weathersby (1994) | ? | 21 |
| | $1.8\times10^{-5}$ | 2400 | Dean and Lange (1999) | ? | 250, 23 |
| | $2.0\times10^{-5}$ | | Yaws and Yang (1992) | ? | 21 |
| | $1.9\times10^{-5}$ | | Abraham et al. (1990) | ? | |
| propane<br>$C_3H_8$<br>[74-98-6]<br><small>ATUOYWHBWRKTHZ-UHFFFAOYSA-N</small> | $1.5\times10^{-5}$ | 2700 | Burkholder et al. (2019) | L | 1 |
| | $1.5\times10^{-5}$ | 2700 | Burkholder et al. (2015) | L | 1 |
| | $1.5\times10^{-5}$ | 2700 | Sander et al. (2011) | L | 1 |
| | $1.5\times10^{-5}$ | 2700 | Sander et al. (2006) | L | 1 |
| | $1.5\times10^{-5}$ | 2800 | Plyasunov and Shock (2000) | L | |
| | $1.5\times10^{-5}$ | 2800 | Abraham and Matteoli (1988) | L | |
| | $1.5\times10^{-5}$ | 2700 | Hayduk (1986) | L | 1 |
| | $1.4\times10^{-5}$ | | Mackay and Shiu (1981) | L | |
| | $1.5\times10^{-5}$ | 2700 | Wilhelm et al. (1977) | L | |
| | $1.6\times10^{-5}$ | 2700 | Chapoy et al. (2004) | M | 1 |
| | $1.5\times10^{-5}$ | 2700 | Reichl (1995) | M | 251 |
| | $9.7\times10^{-6}$ | | Guitart et al. (1989) | M | 14 |
| | $1.4\times10^{-5}$ | 3000 | Ben-Naim and Battino (1985) | M | |
| | $1.4\times10^{-5}$ | | McAuliffe (1966) | M | 225 |
| | $1.6\times10^{-5}$ | 2700 | Wetlaufer et al. (1964) | M | |
| | $1.4\times10^{-5}$ | | McAuliffe (1963) | M | 226 |
| | $1.5\times10^{-5}$ | 2600 | Morrison and Billett (1952) | M | 252 |
| | $1.4\times10^{-5}$ | | Duchowicz et al. (2020) | V | 186 |
| | $1.4\times10^{-5}$ | | HSDB (2015) | V | |
| | $1.4\times10^{-5}$ | | Hine and Mookerjee (1975) | V | |
| | $1.5\times10^{-5}$ | 2600 | Wauchope and Haque (1972) | V | |
| | $1.5\times10^{-5}$ | 2700 | Wauchope and Haque (1972) | V | |





Table A2.1: Alkanes (...continued)

| Substance Formula (Trivial Name) [CAS Registry Number] InChIKey | $H_s^{cp}$ (at $T^{\ominus}$) $\left[\dfrac{\mathrm{mol}}{\mathrm{m^3\,Pa}}\right]$ | $\dfrac{\mathrm{d}\ln H_s^{cp}}{\mathrm{d}(1/T)}$ [K] | Reference | Type | Note |
|---|---|---|---|---|---|
| | $1.3\times10^{-5}$ | | Irmann (1965) | V | |
| | $1.4\times10^{-5}$ | | Yaws (2003) | X | 237 |
| | $1.4\times10^{-5}$ | | Deno and Berkheimer (1960) | C | |
| | $1.7\times10^{-5}$ | | Hayer et al. (2022) | Q | 20 |
| | $4.4\times10^{-4}$ | | Duchowicz et al. (2020) | Q | |
| | $8.7\times10^{-5}$ | | Wang et al. (2017) | Q | 80, 238 |
| | $9.3\times10^{-6}$ | | Wang et al. (2017) | Q | 80, 239 |
| | $1.9\times10^{-5}$ | | Wang et al. (2017) | Q | 80, 240 |
| | $4.1\times10^{-6}$ | | Gharagheizi et al. (2012) | Q | |
| | $1.6\times10^{-5}$ | | Raventos-Duran et al. (2010) | Q | 242, 243 |
| | $1.2\times10^{-5}$ | | Raventos-Duran et al. (2010) | Q | 244 |
| | $1.2\times10^{-5}$ | | Raventos-Duran et al. (2010) | Q | 245 |
| | $2.4\times10^{-5}$ | | Gharagheizi et al. (2010) | Q | 246 |
| | $1.4\times10^{-5}$ | | Hilal et al. (2008) | Q | |
| | $7.2\times10^{-6}$ | | Modarresi et al. (2007) | Q | 67 |
| | | 2900 | Kühne et al. (2005) | Q | |
| | $3.7\times10^{-6}$ | | Modarresi et al. (2005) | Q | 247 |
| | $1.4\times10^{-5}$ | | Yaffe et al. (2003) | Q | 248, 249 |
| | $2.0\times10^{-5}$ | | Yao et al. (2002) | Q | 229 |
| | $1.4\times10^{-5}$ | | English and Carroll (2001) | Q | 230, 231 |
| | $1.5\times10^{-5}$ | | Katritzky et al. (1998) | Q | |
| | $1.7\times10^{-5}$ | | Suzuki et al. (1992) | Q | 232 |
| | $1.6\times10^{-5}$ | | Nirmalakhandan and Speece (1988) | Q | |
| | $1.3\times10^{-5}$ | | Irmann (1965) | Q | |
| | | 2800 | Kühne et al. (2005) | ? | |
| | $1.4\times10^{-5}$ | | Yaws (1999) | ? | 21 |
| | $1.5\times10^{-5}$ | 4500 | Yaws et al. (1999) | ? | 21 |
| | $1.4\times10^{-5}$ | | Yaws and Yang (1992) | ? | 21 |
| | $1.5\times10^{-5}$ | | Abraham et al. (1990) | ? | |
| butane C$_4$H$_{10}$ [106-97-8] IJDNQMDRQITEOD-UHFFFAOYSA-N | $1.2\times10^{-5}$ | 3100 | Burkholder et al. (2019) | L | 253, 1 |
| | $1.2\times10^{-5}$ | 3100 | Burkholder et al. (2015) | L | 254, 1 |
| | $1.2\times10^{-5}$ | 3100 | Sander et al. (2011) | L | 255, 1 |
| | $1.2\times10^{-5}$ | 3100 | Sander et al. (2006) | L | 256, 1 |
| | $1.2\times10^{-5}$ | 3100 | Plyasunov and Shock (2000) | L | |
| | $1.3\times10^{-5}$ | 3100 | Abraham and Matteoli (1988) | L | |
| | $1.2\times10^{-5}$ | 3000 | Hayduk (1986) | L | 1 |
| | $1.0\times10^{-5}$ | | Mackay and Shiu (1981) | L | |
| | $1.2\times10^{-5}$ | 3100 | Wilhelm et al. (1977) | L | |
| | $1.3\times10^{-5}$ | 2300 | Carroll et al. (1997) | M | 1 |
| | $8.0\times10^{-6}$ | | Guitart et al. (1989) | M | 14 |
| | $1.2\times10^{-5}$ | 3200 | Ben-Naim and Battino (1985) | M | |
| | $1.1\times10^{-5}$ | | McAuliffe (1966) | M | 225 |
| | $1.3\times10^{-5}$ | 3200 | Wetlaufer et al. (1964) | M | |
| | $1.1\times10^{-5}$ | | McAuliffe (1963) | M | 226 |
| | $1.1\times10^{-5}$ | 2900 | Morrison and Billett (1952) | M | 257 |
| | $1.0\times10^{-5}$ | | Duchowicz et al. (2020) | V | 186 |
| | $1.0\times10^{-5}$ | | HSDB (2015) | V | |



Table A2.1: Alkanes (... continued)

| Substance Formula (Trivial Name) [CAS Registry Number] InChIKey | $H_s^{cp}$ (at $T^{\ominus}$) $\left[\dfrac{\text{mol}}{\text{m}^3\,\text{Pa}}\right]$ | $\dfrac{\text{d}\ln H_s^{cp}}{\text{d}(1/T)}$ [K] | Reference | Type | Note |
|---|---|---|---|---|---|
| | $1.0\times10^{-5}$ | | Mackay et al. (2006a) | V | |
| | $1.0\times10^{-5}$ | | Mackay et al. (1993) | V | |
| | $9.6\times10^{-6}$ | | Hwang et al. (1992) | V | |
| | $1.1\times10^{-5}$ | | Hine and Mookerjee (1975) | V | |
| | $1.1\times10^{-5}$ | 2900 | Wauchope and Haque (1972) | V | |
| | $1.2\times10^{-5}$ | 3100 | Wauchope and Haque (1972) | V | |
| | $1.2\times10^{-5}$ | | Irmann (1965) | V | |
| | $4.8\times10^{-5}$ | | Butler and Ramchandani (1935) | V | |
| | $1.1\times10^{-5}$ | | Yaws (2003) | X | 258 |
| | $1.1\times10^{-5}$ | | Yaws (2003) | X | 237 |
| | $1.1\times10^{-5}$ | | Deno and Berkheimer (1960) | C | |
| | $1.6\times10^{-5}$ | | Dupeux et al. (2022) | Q | 259 |
| | $1.6\times10^{-5}$ | | Hayer et al. (2022) | Q | 20 |
| | $4.4\times10^{-4}$ | | Duchowicz et al. (2020) | Q | |
| | $7.8\times10^{-5}$ | | Wang et al. (2017) | Q | 80, 238 |
| | $1.1\times10^{-5}$ | | Wang et al. (2017) | Q | 80, 239 |
| | $1.8\times10^{-5}$ | | Wang et al. (2017) | Q | 80, 240 |
| | $1.1\times10^{-5}$ | | Li et al. (2014) | Q | 241 |
| | $4.6\times10^{-6}$ | | Gharagheizi et al. (2012) | Q | |
| | $1.2\times10^{-5}$ | | Raventos-Duran et al. (2010) | Q | 242, 243 |
| | $1.6\times10^{-5}$ | | Raventos-Duran et al. (2010) | Q | 244 |
| | $9.9\times10^{-6}$ | | Raventos-Duran et al. (2010) | Q | 245 |
| | $1.4\times10^{-5}$ | | Gharagheizi et al. (2010) | Q | 246 |
| | $1.2\times10^{-5}$ | | Hilal et al. (2008) | Q | |
| | | 3300 | Kühne et al. (2005) | Q | |
| | $3.5\times10^{-6}$ | | Modarresi et al. (2005) | Q | 247 |
| | $1.1\times10^{-5}$ | | Yaffe et al. (2003) | Q | 248, 249 |
| | $1.6\times10^{-5}$ | | Yao et al. (2002) | Q | 229 |
| | $1.0\times10^{-5}$ | | English and Carroll (2001) | Q | 230, 260 |
| | $1.5\times10^{-5}$ | | Katritzky et al. (1998) | Q | |
| | $1.3\times10^{-5}$ | | Suzuki et al. (1992) | Q | 232 |
| | $1.2\times10^{-5}$ | | Nirmalakhandan and Speece (1988) | Q | |
| | $1.2\times10^{-5}$ | | Irmann (1965) | Q | |
| | | 3300 | Kühne et al. (2005) | ? | |
| | $1.1\times10^{-5}$ | | Yaws (1999) | ? | 21 |
| | $1.2\times10^{-5}$ | 3000 | Yaws et al. (1999) | ? | 21 |
| | $1.1\times10^{-5}$ | | Yaws and Yang (1992) | ? | 21 |
| | $1.2\times10^{-5}$ | | Abraham et al. (1990) | ? | |
| 2-methylpropane HC(CH$_3$)$_3$ (isobutane) [75-28-5] NNPPMTNAJDCUHE-UHFFFAOYSA-N | $9.1\times10^{-6}$ | 2700 | Burkholder et al. (2019) | L | 261, 1 |
| | $9.1\times10^{-6}$ | 2700 | Burkholder et al. (2015) | L | 262, 1 |
| | $9.1\times10^{-6}$ | 2700 | Sander et al. (2011) | L | 263, 1 |
| | $9.1\times10^{-6}$ | 2700 | Sander et al. (2006) | L | 264, 1 |
| | | | Fogg and Sangster (2003) | L | 265 |
| | $9.2\times10^{-6}$ | 2900 | Plyasunov and Shock (2000) | L | |
| | $9.1\times10^{-6}$ | 2700 | Hayduk (1986) | L | 266, 1 |
| | $8.3\times10^{-6}$ | | Mackay and Shiu (1981) | L | |
| | $8.0\times10^{-6}$ | 2700 | Wilhelm et al. (1977) | L | |



Table A2.1: Alkanes (. . . continued)

| Substance Formula (Trivial Name) [CAS Registry Number] InChIKey | $H_s^{cp}$ (at $T^\ominus$) $\left[\dfrac{\mathrm{mol}}{\mathrm{m^3\,Pa}}\right]$ | $\dfrac{\mathrm{d}\ln H_s^{cp}}{\mathrm{d}(1/T)}$ [K] | Reference | Type | Note |
|---|---|---|---|---|---|
| | $1.1\times10^{-4}$ | 5100 | Mohebbi et al. (2012) | M | |
| | $8.5\times10^{-6}$ | | McAuliffe (1966) | M | 225 |
| | $9.9\times10^{-6}$ | 2700 | Wetlaufer et al. (1964) | M | |
| | $8.5\times10^{-6}$ | | McAuliffe (1963) | M | 226 |
| | $8.3\times10^{-6}$ | | Duchowicz et al. (2020) | V | 186 |
| | $8.3\times10^{-6}$ | | HSDB (2015) | V | |
| | $8.3\times10^{-6}$ | | Mackay et al. (2006a) | V | |
| | $8.3\times10^{-6}$ | | Mackay et al. (1993) | V | |
| | $8.4\times10^{-6}$ | | Hine and Mookerjee (1975) | V | |
| | $9.7\times10^{-6}$ | | Irmann (1965) | V | |
| | $8.5\times10^{-6}$ | | Yaws (2003) | X | 237 |
| | $1.3\times10^{-5}$ | | Hayer et al. (2022) | Q | 20 |
| | $1.7\times10^{-4}$ | | Duchowicz et al. (2020) | Q | |
| | $8.3\times10^{-5}$ | | Wang et al. (2017) | Q | 80, 238 |
| | $5.3\times10^{-6}$ | | Wang et al. (2017) | Q | 80, 239 |
| | $2.0\times10^{-5}$ | | Wang et al. (2017) | Q | 80, 240 |
| | $3.1\times10^{-6}$ | | Gharagheizi et al. (2012) | Q | |
| | $1.2\times10^{-5}$ | | Raventos-Duran et al. (2010) | Q | 242, 243 |
| | $6.2\times10^{-6}$ | | Raventos-Duran et al. (2010) | Q | 244 |
| | $9.9\times10^{-6}$ | | Raventos-Duran et al. (2010) | Q | 245 |
| | $1.5\times10^{-5}$ | | Gharagheizi et al. (2010) | Q | 246 |
| | $5.6\times10^{-6}$ | | Hilal et al. (2008) | Q | |
| | $5.8\times10^{-6}$ | | Modarresi et al. (2007) | Q | 67 |
| | | 3300 | Kühne et al. (2005) | Q | |
| | $3.7\times10^{-6}$ | | Modarresi et al. (2005) | Q | 247 |
| | $8.4\times10^{-6}$ | | Yaffe et al. (2003) | Q | 248, 249 |
| | $8.4\times10^{-6}$ | | Yaffe et al. (2003) | Q | 248, 249 |
| | $9.8\times10^{-6}$ | | Yao et al. (2002) | Q | 229, 267 |
| | $1.0\times10^{-5}$ | | English and Carroll (2001) | Q | 230, 231 |
| | $1.5\times10^{-5}$ | | Katritzky et al. (1998) | Q | |
| | $1.1\times10^{-5}$ | | Suzuki et al. (1992) | Q | 232 |
| | $1.0\times10^{-5}$ | | Nirmalakhandan and Speece (1988) | Q | |
| | $1.1\times10^{-5}$ | | Irmann (1965) | Q | |
| | | 2900 | Kühne et al. (2005) | ? | |
| | $8.6\times10^{-6}$ | | Yaws (1999) | ? | 21 |
| | $9.1\times10^{-6}$ | 2700 | Yaws et al. (1999) | ? | 21 |
| | $8.5\times10^{-6}$ | | Yaws and Yang (1992) | ? | 21 |
| | $8.0\times10^{-6}$ | | Abraham et al. (1990) | ? | |
| | $7.9\times10^{-6}$ | | Abraham (1979) | ? | |
| pentane $C_5H_{12}$ [109-66-0] OFBQJSOFQDEBGM-UHFFFAOYSA-N | $7.2\times10^{-6}$ | 3900 | Brockbank (2013) | L | 1, 268 |
| | $8.7\times10^{-6}$ | 3500 | Plyasunov and Shock (2000) | L | |
| | $8.0\times10^{-6}$ | 3400 | Abraham and Matteoli (1988) | L | |
| | $8.0\times10^{-6}$ | | Mackay and Shiu (1981) | L | |
| | $8.7\times10^{-6}$ | 3400 | Jou and Mather (2000) | M | 269, 270 |
| | $8.2\times10^{-6}$ | 3600 | Jönsson et al. (1982) | M | |
| | $7.8\times10^{-6}$ | | Rytting et al. (1978) | M | |
| | $7.8\times10^{-6}$ | | Mackay et al. (2006a) | V | |



Table A2.1: Alkanes (...continued)

| Substance Formula (Trivial Name) [CAS Registry Number] InChIKey | $H_s^{cp}$ (at $T^\ominus$) $\left[\dfrac{\mathrm{mol}}{\mathrm{m}^3\,\mathrm{Pa}}\right]$ | $\dfrac{\mathrm{d}\ln H_s^{cp}}{\mathrm{d}(1/T)}$ [K] | Reference | Type | Note |
|---|---|---|---|---|---|
| | $7.8\times10^{-6}$ | | Mackay et al. (1993) | V | |
| | $8.3\times10^{-6}$ | | Eastcott et al. (1988) | V | |
| | $7.8\times10^{-6}$ | | Amoore and Buttery (1978) | V | |
| | $7.9\times10^{-6}$ | | Hine and Mookerjee (1975) | V | |
| | $8.3\times10^{-6}$ | | McAuliffe (1966) | V | 225 |
| | $8.3\times10^{-6}$ | | McAuliffe (1963) | V | 226 |
| | | 3000 | Gill et al. (1976) | T | |
| | $7.8\times10^{-6}$ | | Yaws (2003) | X | 258 |
| | $7.8\times10^{-6}$ | | Yaws (2003) | X | 237 |
| | $1.3\times10^{-5}$ | | Dupeux et al. (2022) | Q | 259 |
| | $4.9\times10^{-5}$ | | Keshavarz et al. (2022) | Q | |
| | $4.4\times10^{-4}$ | | Duchowicz et al. (2020) | Q | 184 |
| | $6.5\times10^{-5}$ | | Wang et al. (2017) | Q | 80, 238 |
| | $9.3\times10^{-6}$ | | Wang et al. (2017) | Q | 80, 239 |
| | $1.7\times10^{-5}$ | | Wang et al. (2017) | Q | 80, 240 |
| | $4.7\times10^{-6}$ | | Gharagheizi et al. (2012) | Q | |
| | $7.8\times10^{-6}$ | | Raventos-Duran et al. (2010) | Q | 242, 243 |
| | $1.2\times10^{-5}$ | | Raventos-Duran et al. (2010) | Q | 244 |
| | $7.8\times10^{-6}$ | | Raventos-Duran et al. (2010) | Q | 245 |
| | $8.3\times10^{-6}$ | | Gharagheizi et al. (2010) | Q | 246 |
| | $9.9\times10^{-6}$ | | Hilal et al. (2008) | Q | |
| | $6.6\times10^{-6}$ | | Modarresi et al. (2007) | Q | 67 |
| | | 3600 | Kühne et al. (2005) | Q | |
| | $3.1\times10^{-6}$ | | Modarresi et al. (2005) | Q | 247 |
| | $8.4\times10^{-6}$ | | Yaffe et al. (2003) | Q | 248, 249 |
| | $1.5\times10^{-5}$ | | Yao et al. (2002) | Q | 229 |
| | $8.0\times10^{-6}$ | | English and Carroll (2001) | Q | 230, 231 |
| | $1.5\times10^{-5}$ | | Katritzky et al. (1998) | Q | |
| | $1.0\times10^{-5}$ | | Suzuki et al. (1992) | Q | 232 |
| | $9.9\times10^{-6}$ | | Nirmalakhandan and Speece (1988) | Q | |
| | $7.9\times10^{-6}$ | | Duchowicz et al. (2020) | ? | 185, 21 |
| | | 4200 | Kühne et al. (2005) | ? | |
| | $7.8\times10^{-6}$ | | Yaws (1999) | ? | 21 |
| | $5.1\times10^{-6}$ | | Abraham and Weathersby (1994) | ? | 21 |
| | $7.8\times10^{-6}$ | | Yaws and Yang (1992) | ? | 21 |
| | $8.0\times10^{-6}$ | | Abraham et al. (1990) | ? | |
| 2-methylbutane $C_5H_{12}$ (isopentane) [78-78-4] QWTDNUCVQCZILF-UHFFFAOYSA-N | $7.7\times10^{-6}$ | 3000 | Plyasunov and Shock (2000) | L | |
| | $7.2\times10^{-6}$ | | Mackay and Shiu (1981) | L | |
| | $7.0\times10^{-6}$ | | Duchowicz et al. (2020) | V | 186 |
| | $7.0\times10^{-6}$ | | HSDB (2015) | V | |
| | $7.2\times10^{-6}$ | | Mackay et al. (2006a) | V | |
| | $2.1\times10^{-6}$ | | Mackay et al. (1993) | V | |
| | $7.2\times10^{-6}$ | | Eastcott et al. (1988) | V | |
| | $7.2\times10^{-6}$ | | Cabani et al. (1981) | V | |
| | $7.6\times10^{-6}$ | | McAuliffe (1966) | V | 225 |
| | $7.6\times10^{-6}$ | | McAuliffe (1963) | V | 226 |
| | $7.2\times10^{-6}$ | | Yaws (2003) | X | 237 |





Table A2.1: Alkanes (...continued)

| Substance<br>Formula<br>(Trivial Name)<br>[CAS Registry Number]<br>InChIKey | $H_s^{cp}$<br>(at $T^\ominus$)<br>$\left[\dfrac{\text{mol}}{\text{m}^3\,\text{Pa}}\right]$ | $\dfrac{\text{d}\ln H_s^{cp}}{\text{d}(1/T)}$<br><br>[K] | Reference | Type | Note |
|---|---|---|---|---|---|
| | $1.7\times10^{-4}$ | | Duchowicz et al. (2020) | Q | |
| | $7.3\times10^{-5}$ | | Wang et al. (2017) | Q | 80, 238 |
| | $6.0\times10^{-6}$ | | Wang et al. (2017) | Q | 80, 239 |
| | $2.0\times10^{-5}$ | | Wang et al. (2017) | Q | 80, 240 |
| | $3.8\times10^{-6}$ | | Gharagheizi et al. (2012) | Q | |
| | $7.8\times10^{-6}$ | | Raventos-Duran et al. (2010) | Q | 271, 243 |
| | $7.8\times10^{-6}$ | | Raventos-Duran et al. (2010) | Q | 244 |
| | $7.8\times10^{-6}$ | | Raventos-Duran et al. (2010) | Q | 245 |
| | $8.5\times10^{-6}$ | | Gharagheizi et al. (2010) | Q | 246 |
| | $6.4\times10^{-6}$ | | Hilal et al. (2008) | Q | |
| | $7.2\times10^{-6}$ | | Modarresi et al. (2007) | Q | 67 |
| | $3.8\times10^{-6}$ | | Modarresi et al. (2005) | Q | 247 |
| | $5.8\times10^{-6}$ | | Yaffe et al. (2003) | Q | 248, 272 |
| | $8.4\times10^{-6}$ | | Yao et al. (2002) | Q | 229 |
| | $7.9\times10^{-6}$ | | English and Carroll (2001) | Q | 230, 231 |
| | $1.5\times10^{-5}$ | | Katritzky et al. (1998) | Q | |
| | $8.4\times10^{-6}$ | | Nirmalakhandan et al. (1997) | Q | |
| | $7.2\times10^{-6}$ | | Yaws (1999) | ? | 21 |
| | $7.2\times10^{-6}$ | | Yaws and Yang (1992) | ? | 21 |
| dimethylpropane | $6.0\times10^{-6}$ | 3000 | Plyasunov and Shock (2000) | L | |
| $C(CH_3)_4$ | $2.7\times10^{-6}$ | | Mackay and Shiu (1981) | L | |
| (neopentane) | $5.9\times10^{-6}$ | 3300 | Wilhelm et al. (1977) | L | |
| [463-82-1] | $5.5\times10^{-6}$ | 2800 | Shoor et al. (1969) | M | 273 |
| CRSOQBOWXPBRES-UHFFFAOYSA-N | $4.7\times10^{-6}$ | | McAuliffe (1966) | M | |
| | $6.3\times10^{-6}$ | 3200 | Wetlaufer et al. (1964) | M | |
| | $2.7\times10^{-6}$ | | Duchowicz et al. (2020) | V | 186 |
| | $2.7\times10^{-6}$ | | HSDB (2015) | V | |
| | $4.5\times10^{-6}$ | | Mackay et al. (2006a) | V | |
| | $4.5\times10^{-6}$ | | Mackay et al. (1993) | V | |
| | $4.5\times10^{-6}$ | | Hine and Mookerjee (1975) | V | |
| | $4.7\times10^{-6}$ | | Yaws (2003) | X | 237 |
| | $7.5\times10^{-5}$ | | Duchowicz et al. (2020) | Q | |
| | $4.5\times10^{-5}$ | | Wang et al. (2017) | Q | 80, 238 |
| | $2.1\times10^{-6}$ | | Wang et al. (2017) | Q | 80, 239 |
| | $2.5\times10^{-5}$ | | Wang et al. (2017) | Q | 80, 240 |
| | $2.0\times10^{-6}$ | | Gharagheizi et al. (2012) | Q | |
| | $7.8\times10^{-6}$ | | Raventos-Duran et al. (2010) | Q | 242, 243 |
| | $2.5\times10^{-6}$ | | Raventos-Duran et al. (2010) | Q | 244 |
| | $7.8\times10^{-6}$ | | Raventos-Duran et al. (2010) | Q | 245 |
| | $8.3\times10^{-6}$ | | Gharagheizi et al. (2010) | Q | 246 |
| | $2.5\times10^{-6}$ | | Hilal et al. (2008) | Q | |
| | $5.3\times10^{-6}$ | | Modarresi et al. (2007) | Q | 67 |
| | | 3600 | Kühne et al. (2005) | Q | |
| | $3.6\times10^{-6}$ | | Modarresi et al. (2005) | Q | 247 |
| | $4.6\times10^{-6}$ | | Yaffe et al. (2003) | Q | 248, 249 |
| | $3.8\times10^{-6}$ | | Yao et al. (2002) | Q | 229, 267 |
| | $8.0\times10^{-6}$ | | English and Carroll (2001) | Q | 230, 274 |



Table A2.1: Alkanes (...continued)

| Substance Formula (Trivial Name) [CAS Registry Number] InChIKey | $H_s^{cp}$ (at $T^{\ominus}$) $\left[\dfrac{\text{mol}}{\text{m}^3\,\text{Pa}}\right]$ | $\dfrac{\text{d}\ln H_s^{cp}}{\text{d}(1/T)}$ [K] | Reference | Type | Note |
|---|---|---|---|---|---|
| | $1.4\times10^{-5}$ | | Katritzky et al. (1998) | Q | |
| | $7.3\times10^{-6}$ | | Suzuki et al. (1992) | Q | 232 |
| | $6.2\times10^{-6}$ | | Nirmalakhandan and Speece (1988) | Q | |
| | | 3100 | Kühne et al. (2005) | ? | |
| | $4.7\times10^{-6}$ | | Yaws (1999) | ? | 21 |
| | $4.7\times10^{-6}$ | | Yaws and Yang (1992) | ? | 21 |
| | $5.8\times10^{-6}$ | | Abraham et al. (1990) | ? | |
| | $5.9\times10^{-6}$ | | Abraham (1979) | ? | |
| hexane $C_6H_{14}$ [110-54-3] VLKZOEOYAKHREP-UHFFFAOYSA-N | $5.7\times10^{-6}$ | 4400 | Brockbank (2013) | L | 1, 275 |
| | $6.9\times10^{-6}$ | 3800 | Plyasunov and Shock (2000) | L | |
| | $6.1\times10^{-6}$ | 3800 | Abraham and Matteoli (1988) | L | |
| | $5.9\times10^{-6}$ | | Mackay and Shiu (1981) | L | |
| | $6.1\times10^{-6}$ | | Ryu and Park (1999) | M | |
| | $7.4\times10^{-6}$ | | Park et al. (1997) | M | 276 |
| | $2.4\times10^{-4}$ | 8700 | Kolb et al. (1992) | M | 277 |
| | $6.7\times10^{-6}$ | | Guitart et al. (1989) | M | 14 |
| | $9.9\times10^{-6}$ | 7500 | Ashworth et al. (1988) | M | 278 |
| | $6.7\times10^{-6}$ | 4200 | Tsonopoulos and Wilson (1983) | M | 1 |
| | $5.9\times10^{-6}$ | 4000 | Jönsson et al. (1982) | M | |
| | $5.4\times10^{-6}$ | | Rytting et al. (1978) | M | |
| | $5.5\times10^{-6}$ | | Duchowicz et al. (2020) | V | 186 |
| | $5.5\times10^{-6}$ | | HSDB (2015) | V | |
| | $5.5\times10^{-6}$ | | Mackay et al. (2006a) | V | |
| | $5.5\times10^{-6}$ | | Mackay et al. (1993) | V | |
| | $5.5\times10^{-6}$ | | Hwang et al. (1992) | V | |
| | $7.1\times10^{-6}$ | | Eastcott et al. (1988) | V | |
| | $6.1\times10^{-6}$ | | Cabani et al. (1981) | V | |
| | $5.4\times10^{-6}$ | | Hine and Mookerjee (1975) | V | |
| | $5.9\times10^{-6}$ | | McAuliffe (1966) | V | 225 |
| | $5.9\times10^{-6}$ | | McAuliffe (1963) | V | 226 |
| | $6.7\times10^{-6}$ | 3800 | Plyasunov et al. (2001) | T | |
| | | 3800 | Gill et al. (1976) | T | |
| | $7.6\times10^{-6}$ | | Yaws (2003) | X | 258 |
| | $7.6\times10^{-6}$ | | Yaws (2003) | X | 237 |
| | $1.0\times10^{-5}$ | | Dupeux et al. (2022) | Q | 259 |
| | $5.7\times10^{-6}$ | | Hayer et al. (2022) | Q | 20 |
| | $4.4\times10^{-4}$ | | Duchowicz et al. (2020) | Q | |
| | $5.0\times10^{-5}$ | | Wang et al. (2017) | Q | 80, 238 |
| | $6.9\times10^{-6}$ | | Wang et al. (2017) | Q | 80, 239 |
| | $1.6\times10^{-5}$ | | Wang et al. (2017) | Q | 80, 240 |
| | $2.3\times10^{-5}$ | | Li et al. (2014) | Q | 241 |
| | $4.5\times10^{-6}$ | | Gharagheizi et al. (2012) | Q | |
| | $6.2\times10^{-6}$ | | Raventos-Duran et al. (2010) | Q | 242, 243 |
| | $9.9\times10^{-6}$ | | Raventos-Duran et al. (2010) | Q | 244 |
| | $6.2\times10^{-6}$ | | Raventos-Duran et al. (2010) | Q | 245 |
| | $5.2\times10^{-6}$ | | Gharagheizi et al. (2010) | Q | 246 |
| | $7.7\times10^{-6}$ | | Hilal et al. (2008) | Q | |





Table A2.1: Alkanes (...continued)

| Substance Formula (Trivial Name) [CAS Registry Number] InChIKey | $H_s^{cp}$ (at $T^\ominus$) $\left[\dfrac{\text{mol}}{\text{m}^3\,\text{Pa}}\right]$ | $\dfrac{\text{d}\ln H_s^{cp}}{\text{d}(1/T)}$ [K] | Reference | Type | Note |
|---|---|---|---|---|---|
|  | $5.2\times10^{-6}$ |  | Modarresi et al. (2007) | Q | 67 |
|  |  | 4000 | Kühne et al. (2005) | Q |  |
|  | $2.9\times10^{-6}$ |  | Modarresi et al. (2005) | Q | 247 |
|  | $5.8\times10^{-6}$ |  | Yaffe et al. (2003) | Q | 248, 272 |
|  | $6.0\times10^{-6}$ |  | Yao et al. (2002) | Q | 229 |
|  | $6.2\times10^{-6}$ |  | English and Carroll (2001) | Q | 230, 231 |
|  | $1.4\times10^{-5}$ |  | Katritzky et al. (1998) | Q |  |
|  | $1.4\times10^{-5}$ |  | Russell et al. (1992) | Q | 279 |
|  | $7.9\times10^{-6}$ |  | Suzuki et al. (1992) | Q | 232 |
|  | $7.9\times10^{-6}$ |  | Nirmalakhandan and Speece (1988) | Q |  |
|  |  | 4100 | Kühne et al. (2005) | ? |  |
|  | $7.6\times10^{-6}$ |  | Yaws (1999) | ? | 21 |
|  | $3.4\times10^{-6}$ |  | Abraham and Weathersby (1994) | ? | 21 |
|  | $7.6\times10^{-6}$ |  | Yaws and Yang (1992) | ? | 21 |
|  | $6.1\times10^{-6}$ |  | Abraham et al. (1990) | ? |  |
| 2-methylpentane | $6.4\times10^{-6}$ | 4500 | Brockbank (2013) | L | 1 |
| C$_6$H$_{14}$ | $5.7\times10^{-6}$ | 3900 | Plyasunov and Shock (2000) | L |  |
| (isohexane) | $5.9\times10^{-6}$ |  | Mackay and Shiu (1981) | L |  |
| [107-83-5] | $1.3\times10^{-5}$ | 960 | Ashworth et al. (1988) | M | 42, 278 |
| AFABGHUZZDYHJO-UHFFFAOYSA-N | $5.8\times10^{-6}$ |  | Duchowicz et al. (2020) | V | 186 |
|  | $5.8\times10^{-6}$ |  | HSDB (2015) | V |  |
|  | $5.7\times10^{-6}$ |  | Mackay et al. (2006a) | V |  |
|  | $5.7\times10^{-6}$ |  | Mackay et al. (1993) | V |  |
|  | $5.7\times10^{-6}$ |  | Eastcott et al. (1988) | V |  |
|  | $5.7\times10^{-6}$ |  | Hine and Mookerjee (1975) | V |  |
|  | $6.0\times10^{-6}$ |  | McAuliffe (1966) | V | 225 |
|  | $6.0\times10^{-6}$ |  | McAuliffe (1963) | V | 226 |
|  |  |  | Staudinger and Roberts (1996) | R | 280 |
|  | $5.7\times10^{-6}$ |  | Yaws (2003) | X | 237 |
|  | $6.2\times10^{-6}$ |  | Hilal et al. (2008) | C |  |
|  | $1.7\times10^{-4}$ |  | Duchowicz et al. (2020) | Q |  |
|  | $6.0\times10^{-5}$ |  | Wang et al. (2017) | Q | 80, 238 |
|  | $4.5\times10^{-6}$ |  | Wang et al. (2017) | Q | 80, 239 |
|  | $1.9\times10^{-5}$ |  | Wang et al. (2017) | Q | 80, 240 |
|  | $3.5\times10^{-6}$ |  | Gharagheizi et al. (2012) | Q |  |
|  | $6.2\times10^{-6}$ |  | Raventos-Duran et al. (2010) | Q | 242, 243 |
|  | $6.2\times10^{-6}$ |  | Raventos-Duran et al. (2010) | Q | 244 |
|  | $6.2\times10^{-6}$ |  | Raventos-Duran et al. (2010) | Q | 245 |
|  | $5.1\times10^{-6}$ |  | Gharagheizi et al. (2010) | Q | 246 |
|  | $4.8\times10^{-6}$ |  | Hilal et al. (2008) | Q |  |
|  | $5.9\times10^{-6}$ |  | Modarresi et al. (2007) | Q | 67 |
|  |  | 4000 | Kühne et al. (2005) | Q |  |
|  | $3.9\times10^{-6}$ |  | Modarresi et al. (2005) | Q | 247 |
|  | $5.8\times10^{-6}$ |  | Yaffe et al. (2003) | Q | 248, 249 |
|  | $8.0\times10^{-6}$ |  | Yao et al. (2002) | Q | 229 |
|  | $6.1\times10^{-6}$ |  | English and Carroll (2001) | Q | 230, 231 |
|  | $1.4\times10^{-5}$ |  | Katritzky et al. (1998) | Q |  |





Table A2.1: Alkanes (... continued)

| Substance Formula (Trivial Name) [CAS Registry Number] InChIKey | $H_s^{cp}$ (at $T^{\ominus}$) $\left[\dfrac{\mathrm{mol}}{\mathrm{m^3\,Pa}}\right]$ | $\dfrac{\mathrm{d}\ln H_s^{cp}}{\mathrm{d}(1/T)}$ [K] | Reference | Type | Note |
|---|---|---|---|---|---|
| | $7.0\times10^{-6}$ | | Suzuki et al. (1992) | Q | 232 |
| | $6.7\times10^{-6}$ | | Nirmalakhandan and Speece (1988) | Q | |
| | | 4000 | Kühne et al. (2005) | ? | |
| | $5.7\times10^{-6}$ | | Yaws (1999) | ? | 21 |
| | $3.6\times10^{-6}$ | | Abraham and Weathersby (1994) | ? | 21 |
| | $5.7\times10^{-6}$ | | Yaws and Yang (1992) | ? | 21 |
| 3-methylpentane | $5.9\times10^{-6}$ | | Brockbank (2013) | L | |
| $C_6H_{14}$ | $5.9\times10^{-6}$ | | Plyasunov and Shock (2000) | L | |
| [96-14-0] | $5.8\times10^{-6}$ | | Mackay and Shiu (1981) | L | |
| PFEOZHBOMNWTJB-UHFFFAOYSA-N | $5.9\times10^{-6}$ | | Duchowicz et al. (2020) | V | 186 |
| | $5.8\times10^{-6}$ | | HSDB (2015) | V | |
| | $5.9\times10^{-6}$ | | Mackay et al. (2006a) | V | |
| | $5.9\times10^{-6}$ | | Mackay et al. (1993) | V | |
| | $5.9\times10^{-6}$ | | Eastcott et al. (1988) | V | |
| | $5.8\times10^{-6}$ | | Hine and Mookerjee (1975) | V | |
| | $6.0\times10^{-6}$ | | McAuliffe (1966) | V | |
| | $8.2\times10^{-6}$ | | Yaws (2003) | X | 237 |
| | $1.7\times10^{-4}$ | | Duchowicz et al. (2020) | Q | |
| | $6.0\times10^{-5}$ | | Wang et al. (2017) | Q | 80, 238 |
| | $5.8\times10^{-6}$ | | Wang et al. (2017) | Q | 80, 239 |
| | $1.9\times10^{-5}$ | | Wang et al. (2017) | Q | 80, 240 |
| | $4.1\times10^{-6}$ | | Gharagheizi et al. (2012) | Q | |
| | $6.2\times10^{-6}$ | | Raventos-Duran et al. (2010) | Q | 271, 243 |
| | $7.8\times10^{-6}$ | | Raventos-Duran et al. (2010) | Q | 244 |
| | $6.2\times10^{-6}$ | | Raventos-Duran et al. (2010) | Q | 245 |
| | $5.1\times10^{-6}$ | | Gharagheizi et al. (2010) | Q | 246 |
| | $6.2\times10^{-6}$ | | Hilal et al. (2008) | Q | |
| | $7.0\times10^{-6}$ | | Modarresi et al. (2007) | Q | 67 |
| | | 4000 | Kühne et al. (2005) | Q | |
| | $3.1\times10^{-6}$ | | Modarresi et al. (2005) | Q | 247 |
| | $5.8\times10^{-6}$ | | Yaffe et al. (2003) | Q | 248, 249 |
| | $6.7\times10^{-6}$ | | Yao et al. (2002) | Q | 229 |
| | $6.1\times10^{-6}$ | | English and Carroll (2001) | Q | 230, 260 |
| | $1.4\times10^{-5}$ | | Katritzky et al. (1998) | Q | |
| | $7.2\times10^{-6}$ | | Suzuki et al. (1992) | Q | 232 |
| | $7.0\times10^{-6}$ | | Nirmalakhandan and Speece (1988) | Q | |
| | | 4700 | Kühne et al. (2005) | ? | |
| | $8.2\times10^{-6}$ | | Yaws (1999) | ? | 21 |
| | $3.7\times10^{-6}$ | | Abraham and Weathersby (1994) | ? | 21 |
| | $8.8\times10^{-6}$ | | Yaws and Yang (1992) | ? | 21 |



Table A2.1: Alkanes (. . . continued)

| Substance<br>Formula<br>(Trivial Name)<br>[CAS Registry Number]<br>InChIKey | $H_s^{cp}$<br>(at $T^{\ominus}$)<br>$\left[\dfrac{\mathrm{mol}}{\mathrm{m^3\,Pa}}\right]$ | $\dfrac{\mathrm{d}\ln H_s^{cp}}{\mathrm{d}(1/T)}$<br><br>[K] | Reference | Type | Note |
|---|---|---|---|---|---|
| 2,2-dimethylbutane | $5.7\times10^{-6}$ | | Brockbank (2013) | L | |
| $C_6H_{14}$ | $5.6\times10^{-6}$ | 3500 | Plyasunov and Shock (2000) | L | |
| [75-83-2] | $5.8\times10^{-6}$ | | Mackay and Shiu (1981) | L | |
| HNRMPXKDFBEGFZ-UHFFFAOYSA-N | $6.5\times10^{-6}$ | | Duchowicz et al. (2020) | V | 186 |
| | $5.8\times10^{-6}$ | | HSDB (2015) | V | |
| | $5.0\times10^{-6}$ | | Mackay et al. (2006a) | V | |
| | $5.0\times10^{-6}$ | | Mackay et al. (1993) | V | |
| | $5.8\times10^{-6}$ | | Eastcott et al. (1988) | V | |
| | $5.1\times10^{-6}$ | | Hine and Mookerjee (1975) | V | |
| | $5.5\times10^{-6}$ | | McAuliffe (1966) | V | 225 |
| | $5.5\times10^{-6}$ | | McAuliffe (1963) | V | 226 |
| | $6.5\times10^{-6}$ | | Yaws (2003) | X | 237 |
| | $7.5\times10^{-5}$ | | Duchowicz et al. (2020) | Q | |
| | $4.0\times10^{-5}$ | | Wang et al. (2017) | Q | 80, 238 |
| | $2.8\times10^{-6}$ | | Wang et al. (2017) | Q | 80, 239 |
| | $2.5\times10^{-5}$ | | Wang et al. (2017) | Q | 80, 240 |
| | $2.7\times10^{-6}$ | | Gharagheizi et al. (2012) | Q | |
| | $6.2\times10^{-6}$ | | Raventos-Duran et al. (2010) | Q | 271, 243 |
| | $3.1\times10^{-6}$ | | Raventos-Duran et al. (2010) | Q | 244 |
| | $6.2\times10^{-6}$ | | Raventos-Duran et al. (2010) | Q | 245 |
| | $4.9\times10^{-6}$ | | Gharagheizi et al. (2010) | Q | 246 |
| | $3.4\times10^{-6}$ | | Hilal et al. (2008) | Q | |
| | $7.2\times10^{-6}$ | | Modarresi et al. (2007) | Q | 67 |
| | $3.9\times10^{-6}$ | | Modarresi et al. (2005) | Q | 247 |
| | $5.3\times10^{-6}$ | | Yaffe et al. (2003) | Q | 248, 272 |
| | $3.4\times10^{-6}$ | | Yao et al. (2002) | Q | 229 |
| | $6.2\times10^{-6}$ | | English and Carroll (2001) | Q | 230, 231 |
| | $1.3\times10^{-5}$ | | Katritzky et al. (1998) | Q | |
| | $6.0\times10^{-6}$ | | Suzuki et al. (1992) | Q | 232 |
| | $5.3\times10^{-6}$ | | Nirmalakhandan and Speece (1988) | Q | |
| | $6.5\times10^{-6}$ | | Yaws (1999) | ? | 21 |
| | $3.4\times10^{-6}$ | | Abraham and Weathersby (1994) | ? | 21 |
| | $6.5\times10^{-6}$ | | Yaws and Yang (1992) | ? | 21 |
| 2,3-dimethylbutane | $8.0\times10^{-6}$ | 4400 | Brockbank (2013) | L | 1 |
| $C_6H_{14}$ | $7.6\times10^{-6}$ | | Plyasunov and Shock (2000) | L | |
| [79-29-8] | $7.7\times10^{-6}$ | | Mackay and Shiu (1981) | L | |
| ZFFMLCVRJBZUDZ-UHFFFAOYSA-N | $8.4\times10^{-6}$ | | Duchowicz et al. (2020) | V | 186 |
| | $8.2\times10^{-6}$ | | HSDB (2015) | V | |
| | $6.9\times10^{-6}$ | | Mackay et al. (2006a) | V | |
| | $6.9\times10^{-6}$ | | Mackay et al. (1993) | V | |
| | $7.1\times10^{-6}$ | | Eastcott et al. (1988) | V | |
| | $7.6\times10^{-6}$ | | Yaws (2003) | X | 237 |
| | $6.7\times10^{-5}$ | | Duchowicz et al. (2020) | Q | |
| | $6.8\times10^{-5}$ | | Wang et al. (2017) | Q | 80, 238 |
| | $4.9\times10^{-6}$ | | Wang et al. (2017) | Q | 80, 239 |
| | $2.2\times10^{-5}$ | | Wang et al. (2017) | Q | 80, 240 |
| | $3.5\times10^{-6}$ | | Gharagheizi et al. (2012) | Q | |





Table A2.1: Alkanes (…continued)

| Substance Formula (Trivial Name) [CAS Registry Number] InChIKey | $H_s^{cp}$ (at $T^{\ominus}$) $\left[\dfrac{\text{mol}}{\text{m}^3\,\text{Pa}}\right]$ | $\dfrac{\text{d}\ln H_s^{cp}}{\text{d}(1/T)}$ [K] | Reference | Type | Note |
|---|---|---|---|---|---|
| | $6.2\times10^{-6}$ | | Raventos-Duran et al. (2010) | Q | 242, 243 |
| | $6.2\times10^{-6}$ | | Raventos-Duran et al. (2010) | Q | 244 |
| | $6.2\times10^{-6}$ | | Raventos-Duran et al. (2010) | Q | 245 |
| | $5.3\times10^{-6}$ | | Gharagheizi et al. (2010) | Q | 246 |
| | $5.3\times10^{-6}$ | | Hilal et al. (2008) | Q | |
| | $7.6\times10^{-6}$ | | Modarresi et al. (2007) | Q | 67 |
| | | 4000 | Kühne et al. (2005) | Q | |
| | $5.2\times10^{-6}$ | | Modarresi et al. (2005) | Q | 247 |
| | $7.7\times10^{-6}$ | | Yaffe et al. (2003) | Q | 248, 249 |
| | $4.2\times10^{-6}$ | | Yao et al. (2002) | Q | 229 |
| | $6.1\times10^{-6}$ | | English and Carroll (2001) | Q | 230, 274 |
| | $1.3\times10^{-5}$ | | Katritzky et al. (1998) | Q | |
| | $5.8\times10^{-6}$ | | Nirmalakhandan et al. (1997) | Q | |
| | | 4200 | Kühne et al. (2005) | ? | |
| | $7.6\times10^{-6}$ | | Yaws (1999) | ? | 21 |
| | $7.6\times10^{-6}$ | | Yaws and Yang (1992) | ? | 21 |
| heptane C$_7$H$_{16}$ [142-82-5] IMNFDUFMRHMDMM-UHFFFAOYSA-N | $4.4\times10^{-6}$ | 4500 | Brockbank (2013) | L | 1 |
| | $4.9\times10^{-6}$ | 4200 | Plyasunov and Shock (2000) | L | |
| | $4.4\times10^{-6}$ | 4100 | Abraham and Matteoli (1988) | L | |
| | $4.3\times10^{-6}$ | | Mackay and Shiu (1981) | L | |
| | $4.5\times10^{-6}$ | | Ryu and Park (1999) | M | |
| | $5.5\times10^{-6}$ | | Park et al. (1997) | M | 276 |
| | $1.2\times10^{-5}$ | 3700 | Hansen et al. (1993) | M | 281 |
| | $6.0\times10^{-6}$ | | Guitart et al. (1989) | M | 14 |
| | $4.2\times10^{-6}$ | 4700 | Jönsson et al. (1982) | M | |
| | $4.8\times10^{-6}$ | | Rytting et al. (1978) | M | |
| | $4.9\times10^{-6}$ | | Duchowicz et al. (2020) | V | 186 |
| | $5.5\times10^{-6}$ | | HSDB (2015) | V | |
| | $4.8\times10^{-6}$ | | Mackay et al. (2006a) | V | |
| | $4.8\times10^{-6}$ | | Mackay et al. (1993) | V | |
| | $5.0\times10^{-6}$ | | Eastcott et al. (1988) | V | |
| | $4.8\times10^{-6}$ | | Hine and Mookerjee (1975) | V | |
| | $5.2\times10^{-6}$ | | McAuliffe (1966) | V | 225 |
| | $5.2\times10^{-6}$ | | McAuliffe (1963) | V | 226 |
| | $3.7\times10^{-6}$ | | Yaws (2003) | X | 258 |
| | $3.7\times10^{-6}$ | | Yaws (2003) | X | 237 |
| | $8.2\times10^{-6}$ | | Dupeux et al. (2022) | Q | 259 |
| | $1.1\times10^{-5}$ | | Hayer et al. (2022) | Q | 20 |
| | $4.4\times10^{-4}$ | | Duchowicz et al. (2020) | Q | |
| | $4.7\times10^{-5}$ | | Wang et al. (2017) | Q | 80, 238 |
| | $4.9\times10^{-6}$ | | Wang et al. (2017) | Q | 80, 239 |
| | $8.0\times10^{-6}$ | | Wang et al. (2017) | Q | 80, 240 |
| | $4.1\times10^{-6}$ | | Gharagheizi et al. (2012) | Q | |
| | $4.9\times10^{-6}$ | | Raventos-Duran et al. (2010) | Q | 271, 243 |
| | $7.8\times10^{-6}$ | | Raventos-Duran et al. (2010) | Q | 244 |
| | $3.9\times10^{-6}$ | | Raventos-Duran et al. (2010) | Q | 245 |
| | $3.4\times10^{-6}$ | | Gharagheizi et al. (2010) | Q | 246 |



Table A2.1: Alkanes (…continued)

| Substance<br>Formula<br>(Trivial Name)<br>[CAS Registry Number]<br>InChIKey | $H_s^{cp}$<br>(at $T^\ominus$)<br>$\left[\dfrac{\mathrm{mol}}{\mathrm{m^3\,Pa}}\right]$ | $\dfrac{\mathrm{d\ln} H_s^{cp}}{\mathrm{d}(1/T)}$<br><br>[K] | Reference | Type | Note |
|---|---|---|---|---|---|
| | $5.4\times10^{-6}$ | | Hilal et al. (2008) | Q | |
| | $5.3\times10^{-6}$ | | Modarresi et al. (2007) | Q | 67 |
| | | 4300 | Kühne et al. (2005) | Q | |
| | $2.4\times10^{-6}$ | | Modarresi et al. (2005) | Q | 247 |
| | $4.1\times10^{-6}$ | | Yaffe et al. (2003) | Q | 248, 272 |
| | $5.1\times10^{-6}$ | | Yao et al. (2002) | Q | 229 |
| | $1.3\times10^{-5}$ | | Katritzky et al. (1998) | Q | |
| | $6.5\times10^{-6}$ | | Russell et al. (1992) | Q | 279 |
| | $6.0\times10^{-6}$ | | Suzuki et al. (1992) | Q | 232 |
| | $6.2\times10^{-6}$ | | Nirmalakhandan and Speece (1988) | Q | |
| | | 4900 | Kühne et al. (2005) | ? | |
| | $3.7\times10^{-6}$ | | Yaws (1999) | ? | 21 |
| | $2.8\times10^{-6}$ | | Abraham and Weathersby (1994) | ? | 21 |
| | $3.6\times10^{-6}$ | | Yaws and Yang (1992) | ? | 21 |
| | $4.4\times10^{-6}$ | | Abraham et al. (1990) | ? | |
| 2-methylhexane<br>$C_7H_{16}$<br>(isoheptane)<br>[591-76-4]<br>GXDHCNNESPLIKD-UHFFFAOYSA-N | $2.9\times10^{-6}$ | | Brockbank (2013) | L | |
| | $2.9\times10^{-6}$ | | Plyasunov and Shock (2000) | L | |
| | $2.9\times10^{-6}$ | | Mackay and Shiu (1981) | L | |
| | $1.9\times10^{-5}$ | -3600 | Hansen et al. (1993) | M | 281, 282 |
| | $2.9\times10^{-6}$ | | Duchowicz et al. (2020) | V | 186 |
| | $2.9\times10^{-6}$ | | Mackay et al. (2006a) | V | |
| | $2.9\times10^{-6}$ | | Mackay et al. (1993) | V | |
| | $2.9\times10^{-6}$ | | Eastcott et al. (1988) | V | |
| | $2.9\times10^{-6}$ | | Yaws (2003) | X | 237 |
| | $1.7\times10^{-4}$ | | Duchowicz et al. (2020) | Q | |
| | $4.7\times10^{-5}$ | | Wang et al. (2017) | Q | 80, 238 |
| | $3.2\times10^{-6}$ | | Wang et al. (2017) | Q | 80, 239 |
| | $1.7\times10^{-5}$ | | Wang et al. (2017) | Q | 80, 240 |
| | $3.2\times10^{-6}$ | | Gharagheizi et al. (2012) | Q | |
| | $4.9\times10^{-6}$ | | Raventos-Duran et al. (2010) | Q | 271, 243 |
| | $3.9\times10^{-6}$ | | Raventos-Duran et al. (2010) | Q | 244 |
| | $3.9\times10^{-6}$ | | Raventos-Duran et al. (2010) | Q | 245 |
| | $3.3\times10^{-6}$ | | Gharagheizi et al. (2010) | Q | 246 |
| | $3.7\times10^{-6}$ | | Hilal et al. (2008) | Q | |
| | $5.2\times10^{-6}$ | | Modarresi et al. (2007) | Q | 67 |
| | $2.8\times10^{-6}$ | | Modarresi et al. (2005) | Q | 247 |
| | $2.9\times10^{-6}$ | | Yaffe et al. (2003) | Q | 248, 249 |
| | $3.4\times10^{-6}$ | | Yao et al. (2002) | Q | 229 |
| | $1.3\times10^{-5}$ | | Katritzky et al. (1998) | Q | |
| | $5.2\times10^{-6}$ | | Nirmalakhandan et al. (1997) | Q | |
| | $2.9\times10^{-6}$ | | Yaws (1999) | ? | 21 |
| | $2.9\times10^{-6}$ | | Yaws and Yang (1992) | ? | 21 |





Table A2.1: Alkanes (...continued)

| Substance Formula (Trivial Name) [CAS Registry Number] InChIKey | $H_s^{cp}$ (at $T^{\ominus}$) $\left[\dfrac{\text{mol}}{\text{m}^3\,\text{Pa}}\right]$ | $\dfrac{\text{d}\ln H_s^{cp}}{\text{d}(1/T)}$ [K] | Reference | Type | Note |
|---|---|---|---|---|---|
| 3-methylhexane | $3.2\times10^{-6}$ | | Brockbank (2013) | L | |
| $C_7H_{16}$ | $4.5\times10^{-6}$ | | Plyasunov and Shock (2000) | L | |
| [589-34-4] | $4.2\times10^{-6}$ | | Mackay and Shiu (1981) | L | |
| VLJXXKKOSFGPHI-UHFFFAOYSA-N | $6.0\times10^{-6}$ | | Duchowicz et al. (2020) | V | 186 |
| | $4.0\times10^{-6}$ | | Mackay et al. (2006a) | V | |
| | $4.0\times10^{-6}$ | | Mackay et al. (1993) | V | |
| | $3.2\times10^{-6}$ | | Eastcott et al. (1988) | V | |
| | $3.2\times10^{-6}$ | | Yaws (2003) | X | 237 |
| | $1.7\times10^{-4}$ | | Duchowicz et al. (2020) | Q | |
| | $4.7\times10^{-5}$ | | Wang et al. (2017) | Q | 80, 238 |
| | $3.8\times10^{-6}$ | | Wang et al. (2017) | Q | 80, 239 |
| | $1.8\times10^{-5}$ | | Wang et al. (2017) | Q | 80, 240 |
| | $3.6\times10^{-6}$ | | Gharagheizi et al. (2012) | Q | |
| | $3.3\times10^{-6}$ | | Gharagheizi et al. (2010) | Q | 246 |
| | $4.5\times10^{-6}$ | | Hilal et al. (2008) | Q | |
| | $6.1\times10^{-6}$ | | Modarresi et al. (2007) | Q | 67 |
| | $2.7\times10^{-6}$ | | Modarresi et al. (2005) | Q | 247 |
| | $4.1\times10^{-6}$ | | Yaffe et al. (2003) | Q | 248, 249 |
| | $3.4\times10^{-6}$ | | Yao et al. (2002) | Q | 229 |
| | $4.6\times10^{-6}$ | | English and Carroll (2001) | Q | 230, 231 |
| | $1.3\times10^{-5}$ | | Katritzky et al. (1998) | Q | |
| | $5.3\times10^{-6}$ | | Nirmalakhandan et al. (1997) | Q | |
| | $3.2\times10^{-6}$ | | Yaws (1999) | ? | 21 |
| | $2.5\times10^{-6}$ | | Abraham and Weathersby (1994) | ? | 21 |
| | $3.2\times10^{-6}$ | | Yaws and Yang (1992) | ? | 21 |
| 2,2-dimethylpentane | $3.1\times10^{-6}$ | | Plyasunov and Shock (2000) | L | |
| $C_7H_{16}$ | $3.1\times10^{-6}$ | | Mackay and Shiu (1981) | L | |
| [590-35-2] | $3.1\times10^{-6}$ | | Duchowicz et al. (2020) | V | 186 |
| CXOWYJMDMMMMJO-UHFFFAOYSA-N | $3.1\times10^{-6}$ | | Mackay et al. (2006a) | V | |
| | $3.1\times10^{-6}$ | | Mackay et al. (1993) | V | |
| | $3.1\times10^{-6}$ | | Eastcott et al. (1988) | V | |
| | $3.1\times10^{-6}$ | | Yaws (2003) | X | 237 |
| | $7.5\times10^{-5}$ | | Duchowicz et al. (2020) | Q | |
| | $2.4\times10^{-6}$ | | Gharagheizi et al. (2012) | Q | |
| | $4.9\times10^{-6}$ | | Raventos-Duran et al. (2010) | Q | 242, 243 |
| | $2.5\times10^{-6}$ | | Raventos-Duran et al. (2010) | Q | 244 |
| | $3.9\times10^{-6}$ | | Raventos-Duran et al. (2010) | Q | 245 |
| | $3.1\times10^{-6}$ | | Gharagheizi et al. (2010) | Q | 246 |
| | $2.5\times10^{-6}$ | | Hilal et al. (2008) | Q | |
| | $5.2\times10^{-6}$ | | Modarresi et al. (2007) | Q | 67 |
| | $3.1\times10^{-6}$ | | Modarresi et al. (2005) | Q | 247 |
| | $3.1\times10^{-6}$ | | Yaffe et al. (2003) | Q | 248, 249 |
| | $3.7\times10^{-6}$ | | Yao et al. (2002) | Q | 229 |
| | $3.6\times10^{-6}$ | | English and Carroll (2001) | Q | 230, 231 |
| | $1.2\times10^{-5}$ | | Katritzky et al. (1998) | Q | |
| | $4.1\times10^{-6}$ | | Nirmalakhandan et al. (1997) | Q | |
| | $3.1\times10^{-6}$ | | Yaws (1999) | ? | 21 |



Table A2.1: Alkanes (...continued)

| Substance<br>Formula<br>(Trivial Name)<br>[CAS Registry Number]<br>InChIKey | $H_s^{cp}$<br>(at $T^\ominus$)<br>$\left[\dfrac{\mathrm{mol}}{\mathrm{m^3\,Pa}}\right]$ | $\dfrac{\mathrm{d}\ln H_s^{cp}}{\mathrm{d}(1/T)}$<br><br>[K] | Reference | Type | Note |
|---|---|---|---|---|---|
| | $3.1\times10^{-6}$ | | Yaws and Yang (1992) | ? | 21 |
| 2,3-dimethylpentane | $5.7\times10^{-6}$ | | Plyasunov and Shock (2000) | L | |
| $C_7H_{16}$ | $5.7\times10^{-6}$ | | Mackay and Shiu (1981) | L | |
| [565-59-3] | $5.7\times10^{-6}$ | | Duchowicz et al. (2020) | V | 186 |
| WGECXQBGLLYSFP-UHFFFAOYSA-N | $5.7\times10^{-6}$ | | Mackay et al. (1993) | V | |
| | $5.7\times10^{-6}$ | | Eastcott et al. (1988) | V | |
| | $5.7\times10^{-6}$ | | Yaws (2003) | X | 237 |
| | $6.7\times10^{-5}$ | | Duchowicz et al. (2020) | Q | |
| | $3.6\times10^{-6}$ | | Gharagheizi et al. (2012) | Q | |
| | $4.9\times10^{-6}$ | | Raventos-Duran et al. (2010) | Q | 242, 243 |
| | $4.9\times10^{-6}$ | | Raventos-Duran et al. (2010) | Q | 244 |
| | $3.9\times10^{-6}$ | | Raventos-Duran et al. (2010) | Q | 245 |
| | $3.3\times10^{-6}$ | | Gharagheizi et al. (2010) | Q | 246 |
| | $4.8\times10^{-6}$ | | Hilal et al. (2008) | Q | |
| | $7.1\times10^{-6}$ | | Modarresi et al. (2007) | Q | 67 |
| | $5.8\times10^{-6}$ | | Yaffe et al. (2003) | Q | 248, 249 |
| | $4.7\times10^{-6}$ | | English and Carroll (2001) | Q | 230, 231 |
| | $1.2\times10^{-5}$ | | Katritzky et al. (1998) | Q | |
| | $4.7\times10^{-6}$ | | Nirmalakhandan et al. (1997) | Q | |
| | $5.7\times10^{-6}$ | | Yaws (1999) | ? | 21 |
| | $5.7\times10^{-6}$ | | Yaws and Yang (1992) | ? | 21 |
| 2,4-dimethylpentane | $2.9\times10^{-6}$ | | Brockbank (2013) | L | |
| $C_7H_{16}$ | $3.3\times10^{-6}$ | | Plyasunov and Shock (2000) | L | |
| [108-08-7] | $3.3\times10^{-6}$ | | Mackay and Shiu (1981) | L | |
| BZHMBWZPUJHVEE-UHFFFAOYSA-N | $5.2\times10^{-6}$ | | Duchowicz et al. (2020) | V | 186 |
| | $3.1\times10^{-6}$ | | Mackay et al. (2006a) | V | |
| | $3.1\times10^{-6}$ | | Mackay et al. (1993) | V | |
| | $3.4\times10^{-6}$ | | Eastcott et al. (1988) | V | |
| | $3.1\times10^{-6}$ | | Hine and Mookerjee (1975) | V | |
| | $3.1\times10^{-6}$ | | McAuliffe (1966) | V | |
| | $3.3\times10^{-6}$ | | Yaws (2003) | X | 237 |
| | $6.7\times10^{-5}$ | | Duchowicz et al. (2020) | Q | |
| | $2.4\times10^{-6}$ | | Gharagheizi et al. (2012) | Q | |
| | $4.9\times10^{-6}$ | | Raventos-Duran et al. (2010) | Q | 242, 243 |
| | $2.5\times10^{-6}$ | | Raventos-Duran et al. (2010) | Q | 244 |
| | $3.9\times10^{-6}$ | | Raventos-Duran et al. (2010) | Q | 245 |
| | $3.3\times10^{-6}$ | | Gharagheizi et al. (2010) | Q | 246 |
| | $2.2\times10^{-6}$ | | Hilal et al. (2008) | Q | |
| | $5.2\times10^{-6}$ | | Modarresi et al. (2007) | Q | 67 |
| | $3.0\times10^{-6}$ | | Modarresi et al. (2005) | Q | 247 |
| | $5.3\times10^{-6}$ | | Yaffe et al. (2003) | Q | 248, 272 |
| | $4.0\times10^{-6}$ | | Yao et al. (2002) | Q | 229 |
| | $4.7\times10^{-6}$ | | English and Carroll (2001) | Q | 230, 274 |
| | $1.2\times10^{-5}$ | | Katritzky et al. (1998) | Q | |
| | $4.7\times10^{-6}$ | | Suzuki et al. (1992) | Q | 232 |
| | $4.5\times10^{-6}$ | | Nirmalakhandan and Speece (1988) | Q | |





Table A2.1: Alkanes (...continued)

| Substance Formula (Trivial Name) [CAS Registry Number] InChIKey | $H_s^{cp}$ (at $T^\ominus$) $\left[\dfrac{\mathrm{mol}}{\mathrm{m^3\,Pa}}\right]$ | $\dfrac{\mathrm{d}\ln H_s^{cp}}{\mathrm{d}(1/T)}$ [K] | Reference | Type | Note |
|---|---|---|---|---|---|
| | $3.4\times10^{-6}$ | | Yaws (1999) | ? | 21 |
| | $3.3\times10^{-6}$ | | Yaws and Yang (1992) | ? | 21 |
| 3,3-dimethylpentane | $5.6\times10^{-6}$ | 3000 | Brockbank (2013) | L | 1, 283 |
| C$_7$H$_{16}$ | $5.3\times10^{-6}$ | | Plyasunov and Shock (2000) | L | |
| [562-49-2] | $5.4\times10^{-6}$ | | Mackay and Shiu (1981) | L | |
| AEXMKKGTQYQZCS-UHFFFAOYSA-N | $5.4\times10^{-6}$ | | Duchowicz et al. (2020) | V | 186 |
| | $5.4\times10^{-6}$ | | Mackay et al. (2006a) | V | |
| | $5.4\times10^{-6}$ | | Mackay et al. (1993) | V | |
| | $5.4\times10^{-6}$ | | Eastcott et al. (1988) | V | |
| | $5.4\times10^{-6}$ | | Yaws (2003) | X | 237 |
| | $7.5\times10^{-5}$ | | Duchowicz et al. (2020) | Q | |
| | $3.3\times10^{-6}$ | | Gharagheizi et al. (2012) | Q | |
| | $3.1\times10^{-6}$ | | Gharagheizi et al. (2010) | Q | 246 |
| | $4.0\times10^{-6}$ | | Hilal et al. (2008) | Q | |
| | $7.5\times10^{-6}$ | | Modarresi et al. (2007) | Q | 67 |
| | | 4300 | Kühne et al. (2005) | Q | |
| | $3.5\times10^{-6}$ | | Modarresi et al. (2005) | Q | 247 |
| | $5.3\times10^{-6}$ | | Yaffe et al. (2003) | Q | 248, 249 |
| | $2.8\times10^{-6}$ | | Yao et al. (2002) | Q | 229 |
| | $3.6\times10^{-6}$ | | English and Carroll (2001) | Q | 230, 231 |
| | $1.2\times10^{-5}$ | | Katritzky et al. (1998) | Q | |
| | $4.4\times10^{-6}$ | | Nirmalakhandan et al. (1997) | Q | |
| | | 3000 | Kühne et al. (2005) | ? | |
| | $5.4\times10^{-6}$ | | Yaws (1999) | ? | 21 |
| | $5.3\times10^{-6}$ | | Yaws and Yang (1992) | ? | 21 |
| 3-ethylpentane | $3.8\times10^{-6}$ | | Yaws (2003) | X | 237 |
| C$_7$H$_{16}$ | $3.9\times10^{-6}$ | | Gharagheizi et al. (2012) | Q | |
| [617-78-7] | $3.3\times10^{-6}$ | | Gharagheizi et al. (2010) | Q | 246 |
| AORMDLNPRGXHHL-UHFFFAOYSA-N | $5.3\times10^{-6}$ | | Hilal et al. (2008) | Q | |
| | $2.5\times10^{-6}$ | | Modarresi et al. (2005) | Q | 247 |
| | $4.8\times10^{-6}$ | | Yao et al. (2002) | Q | 229, 267 |
| | $3.8\times10^{-6}$ | | Yaws (1999) | ? | 21 |
| | $3.9\times10^{-6}$ | | Yaws and Yang (1992) | ? | 21 |
| 2,2,3-trimethylbutane | $3.2\times10^{-6}$ | | Mackay et al. (2006a) | V | |
| C$_7$H$_{16}$ | $3.2\times10^{-6}$ | | Mackay et al. (1993) | V | |
| [464-06-2] | $4.2\times10^{-6}$ | | Yaws (2003) | X | 237 |
| ZISSAWUMDACLOM-UHFFFAOYSA-N | $2.8\times10^{-6}$ | | Gharagheizi et al. (2012) | Q | |
| | $3.2\times10^{-6}$ | | Gharagheizi et al. (2010) | Q | 246 |
| | $3.3\times10^{-6}$ | | Hilal et al. (2008) | Q | |
| | $7.2\times10^{-6}$ | | Modarresi et al. (2007) | Q | 67 |
| | $3.1\times10^{-6}$ | | Modarresi et al. (2005) | Q | 247 |
| | $3.1\times10^{-6}$ | | Yaffe et al. (2003) | Q | 248, 272 |
| | $1.8\times10^{-6}$ | | Yao et al. (2002) | Q | 229 |
| | $4.2\times10^{-6}$ | | Yaws (1999) | ? | 21 |
| | $4.1\times10^{-6}$ | | Yaws and Yang (1992) | ? | 21 |



Table A2.1: Alkanes (...continued)

| Substance<br>Formula<br>(Trivial Name)<br>[CAS Registry Number]<br>InChIKey | $H_s^{cp}$<br>(at $T^\ominus$)<br><br>$\left[\dfrac{\mathrm{mol}}{\mathrm{m^3\,Pa}}\right]$ | $\dfrac{\mathrm{d}\ln H_s^{cp}}{\mathrm{d}(1/T)}$<br><br>[K] | Reference | Type | Note |
|---|---|---|---|---|---|
| octane | $3.4\times10^{-6}$ | 5300 | Brockbank (2013) | L | 1 |
| $C_8H_{18}$ | $4.0\times10^{-6}$ | 4600 | Plyasunov and Shock (2000) | L | |
| [111-65-9] | $3.1\times10^{-6}$ | 4300 | Abraham and Matteoli (1988) | L | |
| TVMXDCGIABBOFY-UHFFFAOYSA-N | $3.3\times10^{-6}$ | | Mackay and Shiu (1981) | L | |
| | $3.4\times10^{-6}$ | | Ryu and Park (1999) | M | |
| | $3.3\times10^{-6}$ | | Park et al. (1997) | M | 276 |
| | $3.0\times10^{-5}$ | 8000 | Hansen et al. (1993) | M | 281 |
| | $3.1\times10^{-6}$ | 4100 | Heidman et al. (1985) | M | 1 |
| | $2.9\times10^{-6}$ | 5400 | Jönsson et al. (1982) | M | |
| | $3.1\times10^{-6}$ | | Rytting et al. (1978) | M | |
| | $3.1\times10^{-6}$ | | Duchowicz et al. (2020) | V | 186 |
| | $3.1\times10^{-6}$ | | HSDB (2015) | V | |
| | $8.6\times10^{-7}$ | | Abraham and Acree (2007) | V | |
| | $3.2\times10^{-6}$ | | Mackay et al. (2006a) | V | |
| | $3.8\times10^{-6}$ | 4800 | Sarraute et al. (2004) | V | |
| | $3.2\times10^{-6}$ | | Mackay et al. (1993) | V | |
| | $3.0\times10^{-6}$ | | Hwang et al. (1992) | V | |
| | $3.1\times10^{-6}$ | | Meylan and Howard (1991) | V | |
| | $3.2\times10^{-6}$ | | Eastcott et al. (1988) | V | |
| | $3.1\times10^{-6}$ | | Hine and Mookerjee (1975) | V | |
| | $3.1\times10^{-6}$ | | Mackay and Leinonen (1975) | V | |
| | $3.9\times10^{-6}$ | | McAuliffe (1966) | V | 225 |
| | $3.9\times10^{-6}$ | | McAuliffe (1963) | V | 226 |
| | $2.0\times10^{-6}$ | | Yaws (2003) | X | 258 |
| | $2.0\times10^{-6}$ | | Yaws (2003) | X | 237 |
| | $6.8\times10^{-6}$ | | Dupeux et al. (2022) | Q | 259 |
| | $2.8\times10^{-5}$ | | Hayer et al. (2022) | Q | 20 |
| | $4.4\times10^{-4}$ | | Duchowicz et al. (2020) | Q | |
| | $3.6\times10^{-5}$ | | Wang et al. (2017) | Q | 80, 238 |
| | $3.4\times10^{-6}$ | | Wang et al. (2017) | Q | 80, 239 |
| | $4.2\times10^{-6}$ | | Wang et al. (2017) | Q | 80, 240 |
| | $3.1\times10^{-6}$ | | Li et al. (2014) | Q | 241 |
| | $3.5\times10^{-6}$ | | Gharagheizi et al. (2012) | Q | |
| | $3.9\times10^{-6}$ | | Raventos-Duran et al. (2010) | Q | 242, 243 |
| | $4.9\times10^{-6}$ | | Raventos-Duran et al. (2010) | Q | 244 |
| | $3.1\times10^{-6}$ | | Raventos-Duran et al. (2010) | Q | 245 |
| | $2.3\times10^{-6}$ | | Gharagheizi et al. (2010) | Q | 246 |
| | $3.9\times10^{-6}$ | | Hilal et al. (2008) | Q | |
| | $4.6\times10^{-6}$ | | Modarresi et al. (2007) | Q | 67 |
| | | 4700 | Kühne et al. (2005) | Q | |
| | $2.9\times10^{-6}$ | | Yaffe et al. (2003) | Q | 248, 272 |
| | $3.7\times10^{-6}$ | | English and Carroll (2001) | Q | 230, 231 |
| | $1.2\times10^{-5}$ | | Katritzky et al. (1998) | Q | |
| | $2.6\times10^{-6}$ | | Russell et al. (1992) | Q | 279 |
| | $4.6\times10^{-6}$ | | Suzuki et al. (1992) | Q | 232 |
| | $3.3\times10^{-6}$ | | Meylan and Howard (1991) | Q | |
| | $5.0\times10^{-6}$ | | Nirmalakhandan and Speece (1988) | Q | |



Table A2.1: Alkanes (...continued)

| Substance Formula (Trivial Name) [CAS Registry Number] InChIKey | $H_s^{cp}$ (at $T^{\ominus}$) $\left[\dfrac{\mathrm{mol}}{\mathrm{m^3\,Pa}}\right]$ | $\dfrac{\mathrm{d}\ln H_s^{cp}}{\mathrm{d}(1/T)}$ [K] | Reference | Type | Note |
|---|---|---|---|---|---|
| | | 5400 | Kühne et al. (2005) | ? | |
| | $2.0\times10^{-6}$ | | Yaws (1999) | ? | 21 |
| | $2.0\times10^{-6}$ | | Abraham and Weathersby (1994) | ? | 21 |
| | $2.0\times10^{-6}$ | | Yaws and Yang (1992) | ? | 21 |
| | $3.1\times10^{-6}$ | | Abraham et al. (1990) | ? | |
| 2-methylheptane C$_8$H$_{18}$ [592-27-8] JVSWJIKNEAIKJW-UHFFFAOYSA-N | $2.9\times10^{-6}$ | | Mackay et al. (2006a) | V | |
| | $2.9\times10^{-6}$ | | Mackay et al. (1993) | V | |
| | $2.4\times10^{-6}$ | | Yaws (2003) | X | 237 |
| | $2.8\times10^{-6}$ | | Gharagheizi et al. (2012) | Q | |
| | $2.2\times10^{-6}$ | | Gharagheizi et al. (2010) | Q | 246 |
| | $2.7\times10^{-6}$ | | Hilal et al. (2008) | Q | |
| | $2.1\times10^{-6}$ | | Modarresi et al. (2005) | Q | 247 |
| | $3.1\times10^{-6}$ | | Yao et al. (2002) | Q | 229, 267 |
| | $2.4\times10^{-6}$ | | Yaws (1999) | ? | 21 |
| | $2.7\times10^{-6}$ | | Hoff et al. (1993) | ? | 21 |
| | $2.7\times10^{-6}$ | | Yaws and Yang (1992) | ? | 21 |
| 3-methylheptane C$_8$H$_{18}$ [589-81-1] LAIUFBWHERIJIH-UHFFFAOYSA-N | $2.6\times10^{-6}$ | | Brockbank (2013) | L | |
| | $2.7\times10^{-6}$ | | Plyasunov and Shock (2000) | L | |
| | $2.7\times10^{-6}$ | | Mackay and Shiu (1981) | L | |
| | $2.7\times10^{-6}$ | | Duchowicz et al. (2020) | V | 186 |
| | $2.7\times10^{-6}$ | | Eastcott et al. (1988) | V | |
| | $2.6\times10^{-6}$ | | Yaws (2003) | X | 237 |
| | $1.7\times10^{-4}$ | | Duchowicz et al. (2020) | Q | |
| | $3.0\times10^{-6}$ | | Gharagheizi et al. (2012) | Q | |
| | $3.9\times10^{-6}$ | | Raventos-Duran et al. (2010) | Q | 242, 243 |
| | $3.9\times10^{-6}$ | | Raventos-Duran et al. (2010) | Q | 244 |
| | $3.1\times10^{-6}$ | | Raventos-Duran et al. (2010) | Q | 245 |
| | $2.2\times10^{-6}$ | | Gharagheizi et al. (2010) | Q | 246 |
| | $3.3\times10^{-6}$ | | Hilal et al. (2008) | Q | |
| | $5.5\times10^{-6}$ | | Modarresi et al. (2007) | Q | 67 |
| | $2.0\times10^{-6}$ | | Modarresi et al. (2005) | Q | 247 |
| | $2.9\times10^{-6}$ | | Yaffe et al. (2003) | Q | 248, 249 |
| | $3.0\times10^{-6}$ | | Yao et al. (2002) | Q | 229 |
| | $3.6\times10^{-6}$ | | English and Carroll (2001) | Q | 230, 260 |
| | $4.2\times10^{-6}$ | | Nirmalakhandan et al. (1997) | Q | |
| | $2.7\times10^{-6}$ | | Yaws (1999) | ? | 21 |
| | $2.7\times10^{-6}$ | | Yaws and Yang (1992) | ? | 21 |
| 4-methylheptane C$_8$H$_{18}$ [589-53-7] CHBAWFGIXDBEBT-UHFFFAOYSA-N | $2.7\times10^{-6}$ | | Brockbank (2013) | L | |
| | $2.4\times10^{-6}$ | | Yaws (2003) | X | 237 |
| | $2.9\times10^{-6}$ | | Gharagheizi et al. (2012) | Q | |
| | $2.2\times10^{-6}$ | | Gharagheizi et al. (2010) | Q | 246 |
| | $3.0\times10^{-6}$ | | Hilal et al. (2008) | Q | |
| | $2.2\times10^{-6}$ | | Modarresi et al. (2005) | Q | 247 |
| | $2.7\times10^{-6}$ | | Yao et al. (2002) | Q | 229 |
| | $2.4\times10^{-6}$ | | Yaws (1999) | ? | 21 |
| | $2.7\times10^{-6}$ | | Yaws and Yang (1992) | ? | 21 |



Table A2.1: Alkanes (. . . continued)

| Substance Formula (Trivial Name) [CAS Registry Number] InChIKey | $H_s^{cp}$ (at $T^\ominus$) $\left[\dfrac{\text{mol}}{\text{m}^3\,\text{Pa}}\right]$ | $\dfrac{\text{d}\ln H_s^{cp}}{\text{d}(1/T)}$ [K] | Reference | Type | Note |
|---|---|---|---|---|---|
| 2,2-dimethylhexane | $2.4\times10^{-6}$ | 4600 | Brockbank (2013) | L | 1 |
| $C_8H_{18}$ | $2.4\times10^{-6}$ | 4600 | Doháňyosová et al. (2004) | M | 284 |
| [590-73-8] | $2.7\times10^{-7}$ | | Duchowicz et al. (2020) | V | 186 |
| FLTJDUOFAQWHDF-UHFFFAOYSA-N | $2.7\times10^{-6}$ | | Yaws (2003) | X | 237 |
| | $7.5\times10^{-5}$ | | Duchowicz et al. (2020) | Q | |
| | $2.1\times10^{-6}$ | | Gharagheizi et al. (2012) | Q | |
| | $2.0\times10^{-6}$ | | Gharagheizi et al. (2010) | Q | 246 |
| | $1.9\times10^{-6}$ | | Hilal et al. (2008) | Q | |
| | | 4700 | Kühne et al. (2005) | Q | |
| | $4.3\times10^{-6}$ | | Modarresi et al. (2005) | Q | 247 |
| | $1.8\times10^{-6}$ | | Yao et al. (2002) | Q | 229 |
| | | 5100 | Kühne et al. (2005) | ? | |
| | $2.7\times10^{-6}$ | | Yaws (1999) | ? | 21 |
| | $2.9\times10^{-6}$ | | Yaws and Yang (1992) | ? | 21 |
| 2,3-dimethylhexane | $2.4\times10^{-6}$ | | Yaws (2003) | X | 237 |
| $C_8H_{18}$ | $2.9\times10^{-6}$ | | Gharagheizi et al. (2012) | Q | |
| [584-94-1] | $2.2\times10^{-6}$ | | Gharagheizi et al. (2010) | Q | 246 |
| JXPOLSKBTUYKJB-UHFFFAOYSA-N | $3.4\times10^{-6}$ | | Hilal et al. (2008) | Q | |
| | $2.4\times10^{-6}$ | | Yaws (1999) | ? | 21 |
| | $2.6\times10^{-6}$ | | Yaws and Yang (1992) | ? | 21 |
| 2,4-dimethylhexane | $2.6\times10^{-6}$ | | Yaws (2003) | X | 237 |
| $C_8H_{18}$ | $2.3\times10^{-6}$ | | Gharagheizi et al. (2012) | Q | |
| [589-43-5] | $2.2\times10^{-6}$ | | Gharagheizi et al. (2010) | Q | 246 |
| HDGQICNBXPAKLR-UHFFFAOYSA-N | $1.9\times10^{-6}$ | | Hilal et al. (2008) | Q | |
| | $2.6\times10^{-6}$ | | Yaws (1999) | ? | 21 |
| | $2.8\times10^{-6}$ | | Yaws and Yang (1992) | ? | 21 |
| 2,5-dimethylhexane | $3.1\times10^{-6}$ | 4000 | Brockbank (2013) | L | 1, 285 |
| $C_8H_{18}$ | $2.4\times10^{-6}$ | 4000 | Doháňyosová et al. (2004) | M | 286 |
| [592-13-2] | $2.6\times10^{-6}$ | | Yaws (2003) | X | 237 |
| UWNADWZGEHDQAB-UHFFFAOYSA-N | $2.2\times10^{-6}$ | | Gharagheizi et al. (2012) | Q | |
| | $2.2\times10^{-6}$ | | Gharagheizi et al. (2010) | Q | 246 |
| | $1.7\times10^{-6}$ | | Hilal et al. (2008) | Q | |
| | | 4700 | Kühne et al. (2005) | Q | |
| | $2.4\times10^{-6}$ | | Modarresi et al. (2005) | Q | 247 |
| | $2.3\times10^{-6}$ | | Yao et al. (2002) | Q | 229 |
| | | 4700 | Kühne et al. (2005) | ? | |
| | $2.7\times10^{-6}$ | | Yaws (1999) | ? | 21 |
| | $2.9\times10^{-6}$ | | Yaws and Yang (1992) | ? | 21 |
| 3,3-dimethylhexane | $2.4\times10^{-6}$ | | Yaws (2003) | X | 237 |
| $C_8H_{18}$ | $2.7\times10^{-6}$ | | Gharagheizi et al. (2012) | Q | |
| [563-16-6] | $2.0\times10^{-6}$ | | Gharagheizi et al. (2010) | Q | 246 |
| KUMXLFIBWFCMOJ-UHFFFAOYSA-N | $2.9\times10^{-6}$ | | Hilal et al. (2008) | Q | |
| | $1.9\times10^{-6}$ | | Modarresi et al. (2005) | Q | 247 |
| | $1.5\times10^{-6}$ | | Yao et al. (2002) | Q | 229 |
| | $2.4\times10^{-6}$ | | Yaws (1999) | ? | 21 |





Table A2.1: Alkanes (... continued)

| Substance Formula (Trivial Name) [CAS Registry Number] InChIKey | $H_s^{cp}$ (at $T^\ominus$) $\left[\dfrac{\text{mol}}{\text{m}^3\,\text{Pa}}\right]$ | $\dfrac{\text{d}\ln H_s^{cp}}{\text{d}(1/T)}$ [K] | Reference | Type | Note |
|---|---|---|---|---|---|
| | $2.6\times10^{-6}$ | | Yaws and Yang (1992) | ? | 21 |
| 3,4-dimethylhexane $C_8H_{18}$ [583-48-2] RNTWWGNZUXGTAX-UHFFFAOYSA-N | $2.2\times10^{-6}$ | | Yaws (2003) | X | 237 |
| | $2.2\times10^{-6}$ | | Gharagheizi et al. (2010) | Q | 246 |
| | $3.8\times10^{-6}$ | | Hilal et al. (2008) | Q | |
| | $2.3\times10^{-6}$ | | Yaws (1999) | ? | 21 |
| | $2.4\times10^{-6}$ | | Yaws and Yang (1992) | ? | 21 |
| 3-ethylhexane $C_8H_{18}$ [619-99-8] SFRKSDZMZHIISH-UHFFFAOYSA-N | $2.3\times10^{-6}$ | | Yaws (2003) | X | 237 |
| | $2.2\times10^{-6}$ | | Gharagheizi et al. (2010) | Q | 246 |
| | $3.7\times10^{-6}$ | | Hilal et al. (2008) | Q | |
| | $2.3\times10^{-6}$ | | Yaws (1999) | ? | 21 |
| | $2.6\times10^{-6}$ | | Yaws and Yang (1992) | ? | 21 |
| 2,2,3-trimethylpentane $C_8H_{18}$ [564-02-3] XTDQDBVBDLYELW-UHFFFAOYSA-N | $2.4\times10^{-6}$ | | Yaws (2003) | X | 237 |
| | $2.6\times10^{-6}$ | | Gharagheizi et al. (2012) | Q | |
| | $2.0\times10^{-6}$ | | Gharagheizi et al. (2010) | Q | 246 |
| | $2.7\times10^{-6}$ | | Hilal et al. (2008) | Q | |
| | $7.3\times10^{-6}$ | | Modarresi et al. (2007) | Q | 67 |
| | $2.3\times10^{-6}$ | | Modarresi et al. (2005) | Q | 247 |
| | $1.7\times10^{-6}$ | | Yao et al. (2002) | Q | 229 |
| | $2.4\times10^{-6}$ | | Yaws (1999) | ? | 21 |
| | $2.6\times10^{-6}$ | | Yaws and Yang (1992) | ? | 21 |
| 2,2,4-trimethylpentane $C_8H_{18}$ (isooctane) [540-84-1] NHTMVDHEPJAVLT-UHFFFAOYSA-N | $3.0\times10^{-6}$ | | Plyasunov and Shock (2000) | L | |
| | $3.0\times10^{-6}$ | | Mackay and Shiu (1981) | L | |
| | $4.6\times10^{-6}$ | | Guitart et al. (1989) | M | 14 |
| | $3.3\times10^{-6}$ | | Mackay et al. (2006a) | V | |
| | $3.3\times10^{-6}$ | | Mackay et al. (1993) | V | |
| | $3.1\times10^{-6}$ | | Eastcott et al. (1988) | V | |
| | $3.3\times10^{-6}$ | | Hine and Mookerjee (1975) | V | |
| | $3.2\times10^{-6}$ | | Mackay and Leinonen (1975) | V | |
| | $3.6\times10^{-6}$ | | McAuliffe (1966) | V | 225 |
| | $3.6\times10^{-6}$ | | McAuliffe (1963) | V | 226 |
| | $2.8\times10^{-6}$ | | Yaws (2003) | X | 237 |
| | $1.7\times10^{-6}$ | | Gharagheizi et al. (2012) | Q | |
| | $3.9\times10^{-6}$ | | Raventos-Duran et al. (2010) | Q | 242, 243 |
| | $1.2\times10^{-6}$ | | Raventos-Duran et al. (2010) | Q | 244 |
| | $3.1\times10^{-6}$ | | Raventos-Duran et al. (2010) | Q | 245 |
| | $3.3\times10^{-6}$ | | Zhang et al. (2010) | Q | 287, 288 |
| | $1.7\times10^{-6}$ | | Zhang et al. (2010) | Q | 287, 289 |
| | $2.2\times10^{-5}$ | | Zhang et al. (2010) | Q | 287, 290 |
| | $1.6\times10^{-5}$ | | Zhang et al. (2010) | Q | 287, 291 |
| | $2.0\times10^{-6}$ | | Gharagheizi et al. (2010) | Q | 246 |
| | $1.2\times10^{-6}$ | | Hilal et al. (2008) | Q | |
| | $4.8\times10^{-6}$ | | Modarresi et al. (2007) | Q | 67 |
| | | 4700 | Kühne et al. (2005) | Q | |
| | $3.4\times10^{-6}$ | | Modarresi et al. (2005) | Q | 247 |
| | $6.2\times10^{-6}$ | | Yaffe et al. (2003) | Q | 248, 272 |





Table A2.1: Alkanes (...continued)

| Substance Formula (Trivial Name) [CAS Registry Number] InChIKey | $H_s^{cp}$ (at $T^\ominus$) $\left[\dfrac{\text{mol}}{\text{m}^3\,\text{Pa}}\right]$ | $\dfrac{\text{d}\ln H_s^{cp}}{\text{d}(1/T)}$ [K] | Reference | Type | Note |
|---|---|---|---|---|---|
| | $2.0\times10^{-6}$ | | Yao et al. (2002) | Q | 229 |
| | $3.7\times10^{-6}$ | | English and Carroll (2001) | Q | 230, 274 |
| | $3.1\times10^{-6}$ | | Suzuki et al. (1992) | Q | 232 |
| | $2.9\times10^{-6}$ | | Nirmalakhandan and Speece (1988) | Q | |
| | | 4000 | Kühne et al. (2005) | ? | |
| | $2.9\times10^{-6}$ | | Yaws (1999) | ? | 21 |
| | $2.9\times10^{-6}$ | | Yaws and Yang (1992) | ? | 21 |
| 2,3,3-trimethylpentane C$_8$H$_{18}$ [560-21-4] OKVWYBALHQFVFP-UHFFFAOYSA-N | $2.1\times10^{-6}$ | | Yaws (2003) | X | 237 |
| | $3.2\times10^{-6}$ | | Gharagheizi et al. (2012) | Q | |
| | $2.0\times10^{-6}$ | | Gharagheizi et al. (2010) | Q | 246 |
| | $3.6\times10^{-6}$ | | Hilal et al. (2008) | Q | |
| | $2.0\times10^{-6}$ | | Modarresi et al. (2005) | Q | 247 |
| | $1.6\times10^{-6}$ | | Yao et al. (2002) | Q | 229 |
| | $2.2\times10^{-6}$ | | Yaws (1999) | ? | 21 |
| | $2.4\times10^{-6}$ | | Yaws and Yang (1992) | ? | 21 |
| 2,3,4-trimethylpentane C$_8$H$_{18}$ [565-75-3] RLPGDEORIPLBNF-UHFFFAOYSA-N | $3.3\times10^{-6}$ | | Brockbank (2013) | L | |
| | $4.3\times10^{-6}$ | | Plyasunov and Shock (2000) | L | |
| | $5.3\times10^{-6}$ | | Mackay and Shiu (1981) | L | |
| | | | Mackay et al. (2006a) | V | 292 |
| | $4.9\times10^{-6}$ | | Mackay et al. (1993) | V | |
| | $5.6\times10^{-6}$ | | Eastcott et al. (1988) | V | |
| | $5.6\times10^{-6}$ | | Yaws (2003) | X | 237 |
| | $2.9\times10^{-6}$ | | Gharagheizi et al. (2012) | Q | |
| | $3.9\times10^{-6}$ | | Raventos-Duran et al. (2010) | Q | 242, 243 |
| | $3.1\times10^{-6}$ | | Raventos-Duran et al. (2010) | Q | 244 |
| | $3.1\times10^{-6}$ | | Raventos-Duran et al. (2010) | Q | 245 |
| | $2.2\times10^{-6}$ | | Gharagheizi et al. (2010) | Q | 246 |
| | $3.1\times10^{-6}$ | | Hilal et al. (2008) | Q | |
| | $7.0\times10^{-6}$ | | Modarresi et al. (2007) | Q | 67 |
| | | 4700 | Kühne et al. (2005) | Q | |
| | $2.0\times10^{-6}$ | | Modarresi et al. (2005) | Q | 247 |
| | $5.3\times10^{-6}$ | | Yaffe et al. (2003) | Q | 248, 249 |
| | $2.1\times10^{-6}$ | | Yao et al. (2002) | Q | 229 |
| | $3.7\times10^{-6}$ | | English and Carroll (2001) | Q | 230, 231 |
| | $1.1\times10^{-5}$ | | Katritzky et al. (1998) | Q | |
| | $3.2\times10^{-6}$ | | Nirmalakhandan et al. (1997) | Q | |
| | | 4900 | Kühne et al. (2005) | ? | |
| | $5.6\times10^{-6}$ | | Yaws (1999) | ? | 21 |
| | $5.6\times10^{-6}$ | | Yaws and Yang (1992) | ? | 21 |
| 3-ethyl-2-methylpentane C$_8$H$_{18}$ [609-26-7] DUPUVYJQZSLSJB-UHFFFAOYSA-N | $2.3\times10^{-6}$ | | Yaws (2003) | X | 237 |
| | $3.1\times10^{-6}$ | | Gharagheizi et al. (2012) | Q | |
| | $2.2\times10^{-6}$ | | Gharagheizi et al. (2010) | Q | 246 |
| | $3.6\times10^{-6}$ | | Hilal et al. (2008) | Q | |
| | $2.2\times10^{-6}$ | | Modarresi et al. (2005) | Q | 247 |
| | $2.5\times10^{-6}$ | | Yao et al. (2002) | Q | 229 |
| | $2.3\times10^{-6}$ | | Yaws (1999) | ? | 21 |





Table A2.1: Alkanes (. . . continued)

| Substance Formula (Trivial Name) [CAS Registry Number] InChIKey | $H_s^{cp}$ (at $T^{\ominus}$) $\left[\dfrac{\text{mol}}{\text{m}^3\,\text{Pa}}\right]$ | $\dfrac{\text{d}\ln H_s^{cp}}{\text{d}(1/T)}$ [K] | Reference | Type | Note |
|---|---|---|---|---|---|
| | $2.6\times10^{-6}$ | | Yaws and Yang (1992) | ? | 21 |
| 3-ethyl-3-methylpentane | $2.1\times10^{-6}$ | | Yaws (2003) | X | 237 |
| $C_8H_{18}$ | $3.5\times10^{-6}$ | | Gharagheizi et al. (2012) | Q | |
| [1067-08-9] | $2.0\times10^{-6}$ | | Gharagheizi et al. (2010) | Q | 246 |
| GIEZWIDCIFCQPS-UHFFFAOYSA-N | $4.5\times10^{-6}$ | | Hilal et al. (2008) | Q | |
| | $2.0\times10^{-6}$ | | Modarresi et al. (2005) | Q | 247 |
| | $2.2\times10^{-6}$ | | Yao et al. (2002) | Q | 229 |
| | $2.1\times10^{-6}$ | | Yaws (1999) | ? | 21 |
| | $2.3\times10^{-6}$ | | Yaws and Yang (1992) | ? | 21 |
| 2,2,3,3-tetramethylbutane | $3.3\times10^{-6}$ | | Yaws (2003) | X | 237 |
| $C_8H_{18}$ | $2.6\times10^{-6}$ | | Gharagheizi et al. (2012) | Q | |
| [594-82-1] | $2.0\times10^{-6}$ | | Gharagheizi et al. (2010) | Q | 246 |
| OMMLUKLXGSRPHK-UHFFFAOYSA-N | $3.4\times10^{-6}$ | | Hilal et al. (2008) | Q | |
| | $2.6\times10^{-6}$ | | Yaws and Yang (1992) | ? | 21 |
| nonane | $2.3\times10^{-6}$ | 6600 | Brockbank (2013) | L | 1, 293 |
| $C_9H_{20}$ | $2.1\times10^{-6}$ | | Plyasunov and Shock (2000) | L | |
| [111-84-2] | $2.0\times10^{-6}$ | | Mackay and Shiu (1981) | L | |
| BKIMMITUMNQMOS-UHFFFAOYSA-N | $2.2\times10^{-6}$ | | Ryu and Park (1999) | M | |
| | $1.9\times10^{-6}$ | | Park et al. (1997) | M | 276 |
| | $2.3\times10^{-5}$ | 190 | Ashworth et al. (1988) | M | 42, 278 |
| | $1.8\times10^{-6}$ | 7300 | Jönsson et al. (1982) | M | |
| | $2.9\times10^{-6}$ | | Duchowicz et al. (2020) | V | 186 |
| | $2.9\times10^{-6}$ | | HSDB (2015) | V | |
| | $3.0\times10^{-6}$ | | Mackay et al. (2006a) | V | |
| | $3.0\times10^{-6}$ | | Mackay et al. (1993) | V | |
| | $1.7\times10^{-6}$ | | Eastcott et al. (1988) | V | |
| | $2.0\times10^{-6}$ | | Abraham (1984) | V | |
| | $1.6\times10^{-6}$ | | Yaws (2003) | X | 237 |
| | $4.4\times10^{-4}$ | | Duchowicz et al. (2020) | Q | |
| | $2.9\times10^{-5}$ | | Wang et al. (2017) | Q | 80, 238 |
| | $2.5\times10^{-6}$ | | Wang et al. (2017) | Q | 80, 239 |
| | $2.3\times10^{-5}$ | | Wang et al. (2017) | Q | 80, 240 |
| | $3.0\times10^{-6}$ | | Gharagheizi et al. (2012) | Q | |
| | $2.5\times10^{-6}$ | | Raventos-Duran et al. (2010) | Q | 271, 243 |
| | $3.9\times10^{-6}$ | | Raventos-Duran et al. (2010) | Q | 244 |
| | $2.5\times10^{-6}$ | | Raventos-Duran et al. (2010) | Q | 245 |
| | $1.8\times10^{-6}$ | | Gharagheizi et al. (2010) | Q | 246 |
| | $3.0\times10^{-6}$ | | Hilal et al. (2008) | Q | |
| | $4.3\times10^{-6}$ | | Modarresi et al. (2007) | Q | 67 |
| | | 5000 | Kühne et al. (2005) | Q | |
| | $1.5\times10^{-6}$ | | Modarresi et al. (2005) | Q | 247 |
| | $2.0\times10^{-6}$ | | Yaffe et al. (2003) | Q | 248, 249 |
| | $4.6\times10^{-6}$ | | Yao et al. (2002) | Q | 229 |
| | $1.1\times10^{-5}$ | | Katritzky et al. (1998) | Q | |
| | $3.8\times10^{-6}$ | | Nirmalakhandan et al. (1997) | Q | |
| | | 4100 | Kühne et al. (2005) | ? | |



Table A2.1: Alkanes (. . . continued)

| Substance Formula (Trivial Name) [CAS Registry Number] InChIKey | $H_s^{cp}$ (at $T^{\ominus}$) $\left[\dfrac{\mathrm{mol}}{\mathrm{m^3\,Pa}}\right]$ | $\dfrac{\mathrm{d}\ln H_s^{cp}}{\mathrm{d}(1/T)}$ [K] | Reference | Type | Note |
|---|---|---|---|---|---|
| | $1.6{\times}10^{-6}$ | | Yaws (1999) | ? | 21 |
| | $1.7{\times}10^{-6}$ | | Yaws and Yang (1992) | ? | 21 |
| 2-methyloctane | $1.6{\times}10^{-6}$ | | Yaws (2003) | X | 237 |
| $C_9H_{20}$ | $2.4{\times}10^{-6}$ | | Gharagheizi et al. (2012) | Q | |
| [3221-61-2] | $1.6{\times}10^{-6}$ | | Gharagheizi et al. (2010) | Q | 246 |
| ZUBZATZOEPUUQF-UHFFFAOYSA-N | $1.9{\times}10^{-6}$ | | Hilal et al. (2008) | Q | |
| | $2.1{\times}10^{-6}$ | | Modarresi et al. (2005) | Q | 247 |
| | $3.3{\times}10^{-6}$ | | Yao et al. (2002) | Q | 229 |
| | $1.6{\times}10^{-6}$ | | Yaws (1999) | ? | 21 |
| | $2.1{\times}10^{-6}$ | | Yaws and Yang (1992) | ? | 21 |
| 3-methyloctane | $1.5{\times}10^{-6}$ | | Yaws (2003) | X | 237 |
| $C_9H_{20}$ | $2.6{\times}10^{-6}$ | | Gharagheizi et al. (2012) | Q | |
| [2216-33-3] | $1.6{\times}10^{-6}$ | | Gharagheizi et al. (2010) | Q | 246 |
| SEEOMASXHIJCDV-UHFFFAOYSA-N | $2.4{\times}10^{-6}$ | | Hilal et al. (2008) | Q | |
| | $1.6{\times}10^{-6}$ | | Modarresi et al. (2005) | Q | 247 |
| | $2.8{\times}10^{-6}$ | | Yao et al. (2002) | Q | 229 |
| | $1.5{\times}10^{-6}$ | | Yaws (1999) | ? | 21 |
| | $1.9{\times}10^{-6}$ | | Yaws and Yang (1992) | ? | 21 |
| 4-methyloctane | $9.7{\times}10^{-7}$ | | Plyasunov and Shock (2000) | L | |
| $C_9H_{20}$ | $1.0{\times}10^{-6}$ | | Mackay and Shiu (1981) | L | |
| [2216-34-4] | $9.9{\times}10^{-7}$ | | Duchowicz et al. (2020) | V | 186 |
| DOGIHOCMZJUJNR-UHFFFAOYSA-N | $9.9{\times}10^{-7}$ | | Eastcott et al. (1988) | V | |
| | $9.8{\times}10^{-7}$ | | Yaws (2003) | X | 237 |
| | $1.7{\times}10^{-4}$ | | Duchowicz et al. (2020) | Q | |
| | $2.4{\times}10^{-6}$ | | Gharagheizi et al. (2012) | Q | |
| | $2.5{\times}10^{-6}$ | | Raventos-Duran et al. (2010) | Q | 242, 243 |
| | $3.1{\times}10^{-6}$ | | Raventos-Duran et al. (2010) | Q | 244 |
| | $2.5{\times}10^{-6}$ | | Raventos-Duran et al. (2010) | Q | 245 |
| | $1.6{\times}10^{-6}$ | | Gharagheizi et al. (2010) | Q | 246 |
| | $2.3{\times}10^{-6}$ | | Hilal et al. (2008) | Q | |
| | $4.9{\times}10^{-6}$ | | Modarresi et al. (2007) | Q | 67 |
| | $1.6{\times}10^{-6}$ | | Modarresi et al. (2005) | Q | 247 |
| | $2.9{\times}10^{-6}$ | | Yao et al. (2002) | Q | 229 |
| | $9.8{\times}10^{-7}$ | | Yaws (1999) | ? | 21 |
| | $9.9{\times}10^{-7}$ | | Yaws and Yang (1992) | ? | 21 |
| 2,3-dimethylheptane | $1.4{\times}10^{-6}$ | | Yaws (2003) | X | 237 |
| $C_9H_{20}$ | $2.4{\times}10^{-6}$ | | Gharagheizi et al. (2012) | Q | |
| [3074-71-3] | $1.5{\times}10^{-6}$ | | Gharagheizi et al. (2010) | Q | 246 |
| WBRFDUJXCLCKPX-UHFFFAOYSA-N | $2.4{\times}10^{-6}$ | | Hilal et al. (2008) | Q | |
| | $2.4{\times}10^{-6}$ | | Modarresi et al. (2005) | Q | 247 |
| | $2.0{\times}10^{-6}$ | | Yao et al. (2002) | Q | 229, 267 |
| | $1.5{\times}10^{-6}$ | | Yaws (1999) | ? | 21 |
| | $1.9{\times}10^{-6}$ | | Yaws and Yang (1992) | ? | 21 |



Table A2.1: Alkanes (... continued)

| Substance Formula (Trivial Name) [CAS Registry Number] InChIKey | $H_s^{cp}$ (at $T^{\ominus}$) $\left[\dfrac{\text{mol}}{\text{m}^3\,\text{Pa}}\right]$ | $\dfrac{\text{d}\ln H_s^{cp}}{\text{d}(1/T)}$ [K] | Reference | Type | Note |
|---|---|---|---|---|---|
| 2,2-dimethylheptane | $1.7\times10^{-6}$ | | Yaws (2003) | X | 237 |
| $C_9H_{20}$ | $1.8\times10^{-6}$ | | Gharagheizi et al. (2012) | Q | |
| [1071-26-7] | $1.4\times10^{-6}$ | | Gharagheizi et al. (2010) | Q | 246 |
| PSABUFWDVWCFDP-UHFFFAOYSA-N | $1.4\times10^{-6}$ | | Hilal et al. (2008) | Q | |
| | $1.6\times10^{-6}$ | | Modarresi et al. (2005) | Q | 247 |
| | $2.0\times10^{-6}$ | | Yao et al. (2002) | Q | 229 |
| | $1.7\times10^{-6}$ | | Yaws (1999) | ? | 21 |
| | $2.1\times10^{-6}$ | | Yaws and Yang (1992) | ? | 21 |
| 2,4-dimethylheptane | $1.7\times10^{-6}$ | | Yaws (2003) | X | 237 |
| $C_9H_{20}$ | $1.8\times10^{-6}$ | | Gharagheizi et al. (2012) | Q | |
| [2213-23-2] | $1.5\times10^{-6}$ | | Gharagheizi et al. (2010) | Q | 246 |
| AUKVIBNBLXQNIZ-UHFFFAOYSA-N | $1.4\times10^{-6}$ | | Hilal et al. (2008) | Q | |
| | $1.6\times10^{-6}$ | | Modarresi et al. (2005) | Q | 247 |
| | $1.8\times10^{-6}$ | | Yao et al. (2002) | Q | 229 |
| | $1.7\times10^{-6}$ | | Yaws (1999) | ? | 21 |
| | $2.1\times10^{-6}$ | | Yaws and Yang (1992) | ? | 21 |
| 2,5-dimethylheptane | $1.6\times10^{-6}$ | | Yaws (2003) | X | 237 |
| $C_9H_{20}$ | $2.0\times10^{-6}$ | | Gharagheizi et al. (2012) | Q | |
| [2216-30-0] | $1.5\times10^{-6}$ | | Gharagheizi et al. (2010) | Q | 246 |
| HQZHQNKZOYIKQC-UHFFFAOYSA-N | $1.5\times10^{-6}$ | | Hilal et al. (2008) | Q | |
| | $2.0\times10^{-6}$ | | Yao et al. (2002) | Q | 229 |
| | $1.6\times10^{-6}$ | | Yaws (1999) | ? | 21 |
| | $2.0\times10^{-6}$ | | Yaws and Yang (1992) | ? | 21 |
| 2,6-dimethylheptane | $1.7\times10^{-6}$ | | Yaws (2003) | X | 237 |
| $C_9H_{20}$ | $1.9\times10^{-6}$ | | Gharagheizi et al. (2012) | Q | |
| [1072-05-5] | $1.5\times10^{-6}$ | | Gharagheizi et al. (2010) | Q | 246 |
| KBPCCVWUMVGXGF-UHFFFAOYSA-N | $1.2\times10^{-6}$ | | Hilal et al. (2008) | Q | |
| | $1.6\times10^{-6}$ | | Modarresi et al. (2005) | Q | 247 |
| | $2.2\times10^{-6}$ | | Yao et al. (2002) | Q | 229 |
| | $1.7\times10^{-6}$ | | Yaws (1999) | ? | 21 |
| | $2.1\times10^{-6}$ | | Yaws and Yang (1992) | ? | 21 |
| 3,3-dimethylheptane | $2.3\times10^{-6}$ | | Hilal et al. (2008) | Q | |
| $C_9H_{20}$ | $1.9\times10^{-6}$ | | Yaws and Yang (1992) | ? | 21 |
| [4032-86-4] | | | | | |
| BVAKDOXCVSMKHE-UHFFFAOYSA-N | | | | | |
| 3,4-dimethylheptane | $1.4\times10^{-6}$ | | Yaws (2003) | X | 237 |
| $C_9H_{20}$ | $2.5\times10^{-6}$ | | Gharagheizi et al. (2012) | Q | |
| [922-28-1] | $1.5\times10^{-6}$ | | Gharagheizi et al. (2010) | Q | 246 |
| MAKRYGRRIKSDES-UHFFFAOYSA-N | $2.6\times10^{-6}$ | | Hilal et al. (2008) | Q | |
| | $1.7\times10^{-6}$ | | Modarresi et al. (2005) | Q | 247 |
| | $1.6\times10^{-6}$ | | Yao et al. (2002) | Q | 229 |
| | $1.4\times10^{-6}$ | | Yaws (1999) | ? | 21 |
| | $1.8\times10^{-6}$ | | Yaws and Yang (1992) | ? | 21 |



Table A2.1: Alkanes (...continued)

| Substance Formula (Trivial Name) [CAS Registry Number] InChIKey | $H_s^{cp}$ (at $T^{\ominus}$) $\left[\dfrac{\mathrm{mol}}{\mathrm{m}^3\,\mathrm{Pa}}\right]$ | $\dfrac{\mathrm{d}\ln H_s^{cp}}{\mathrm{d}(1/T)}$ [K] | Reference | Type | Note |
|---|---|---|---|---|---|
| 3,5-dimethylheptane | $1.6\times10^{-6}$ | | Yaws (2003) | X | 237 |
| $C_9H_{20}$ | $2.1\times10^{-6}$ | | Gharagheizi et al. (2012) | Q | |
| [926-82-9] | $1.5\times10^{-6}$ | | Gharagheizi et al. (2010) | Q | 246 |
| DZJTZGHZAWTWGA-UHFFFAOYSA-N | $1.5\times10^{-6}$ | | Hilal et al. (2008) | Q | |
| | $1.7\times10^{-6}$ | | Modarresi et al. (2005) | Q | 247 |
| | $1.6\times10^{-6}$ | | Yao et al. (2002) | Q | 229 |
| | $1.6\times10^{-6}$ | | Yaws (1999) | ? | 21 |
| | $2.0\times10^{-6}$ | | Yaws and Yang (1992) | ? | 21 |
| 4,4-dimethylheptane | $1.5\times10^{-6}$ | | Yaws (2003) | X | 237 |
| $C_9H_{20}$ | $2.1\times10^{-6}$ | | Gharagheizi et al. (2012) | Q | |
| [1068-19-5] | $1.4\times10^{-6}$ | | Gharagheizi et al. (2010) | Q | 246 |
| WSOKFYJGNBQDPW-UHFFFAOYSA-N | $2.1\times10^{-6}$ | | Hilal et al. (2008) | Q | |
| | $1.5\times10^{-6}$ | | Modarresi et al. (2005) | Q | 247 |
| | $1.5\times10^{-6}$ | | Yao et al. (2002) | Q | 229 |
| | $1.5\times10^{-6}$ | | Yaws (1999) | ? | 21 |
| | $1.9\times10^{-6}$ | | Yaws and Yang (1992) | ? | 21 |
| 3-ethylheptane | $1.4\times10^{-6}$ | | Yaws (2003) | X | 237 |
| $C_9H_{20}$ | $2.6\times10^{-6}$ | | Gharagheizi et al. (2012) | Q | |
| [15869-80-4] | $1.6\times10^{-6}$ | | Gharagheizi et al. (2010) | Q | 246 |
| PSVQKOKKLWHNRP-UHFFFAOYSA-N | $2.6\times10^{-6}$ | | Hilal et al. (2008) | Q | |
| | $1.6\times10^{-6}$ | | Modarresi et al. (2005) | Q | 247 |
| | $2.4\times10^{-6}$ | | Yao et al. (2002) | Q | 229 |
| | $1.4\times10^{-6}$ | | Yaws (1999) | ? | 21 |
| | $1.9\times10^{-6}$ | | Yaws and Yang (1992) | ? | 21 |
| 4-ethylheptane | $1.5\times10^{-6}$ | | Yaws (2003) | X | 237 |
| $C_9H_{20}$ | $2.3\times10^{-6}$ | | Gharagheizi et al. (2012) | Q | |
| [2216-32-2] | $1.6\times10^{-6}$ | | Gharagheizi et al. (2010) | Q | 246 |
| XMROPFQWHHUFFS-UHFFFAOYSA-N | $2.5\times10^{-6}$ | | Hilal et al. (2008) | Q | |
| | $1.5\times10^{-6}$ | | Modarresi et al. (2005) | Q | 247 |
| | $2.8\times10^{-6}$ | | Yao et al. (2002) | Q | 229 |
| | $1.5\times10^{-6}$ | | Yaws (1999) | ? | 21 |
| | $1.9\times10^{-6}$ | | Yaws and Yang (1992) | ? | 21 |
| 2,2,3-trimethylhexane | $1.5\times10^{-6}$ | | Yaws (2003) | X | 237 |
| $C_9H_{20}$ | $2.1\times10^{-6}$ | | Gharagheizi et al. (2012) | Q | |
| [16747-25-4] | $1.4\times10^{-6}$ | | Gharagheizi et al. (2010) | Q | 246 |
| CBVFSZDQEHBJEQ-UHFFFAOYSA-N | $1.9\times10^{-6}$ | | Hilal et al. (2008) | Q | |
| | $1.6\times10^{-6}$ | | Modarresi et al. (2005) | Q | 247 |
| | $9.0\times10^{-7}$ | | Yao et al. (2002) | Q | 229 |
| | $1.5\times10^{-6}$ | | Yaws (1999) | ? | 21 |
| | $1.9\times10^{-6}$ | | Yaws and Yang (1992) | ? | 21 |





Table A2.1: Alkanes (. . . continued)

| Substance<br>Formula<br>(Trivial Name)<br>[CAS Registry Number]<br>InChIKey | $H_s^{cp}$ (at $T^{\ominus}$) $\left[\dfrac{\mathrm{mol}}{\mathrm{m^3\,Pa}}\right]$ | $\dfrac{\mathrm{d}\ln H_s^{cp}}{\mathrm{d}(1/T)}$ [K] | Reference | Type | Note |
|---|---|---|---|---|---|
| 2,2,4-trimethylhexane | $1.7\times10^{-6}$ | | Yaws (2003) | X | 237 |
| C$_9$H$_{20}$ | $1.6\times10^{-6}$ | | Gharagheizi et al. (2012) | Q | |
| [16747-26-5] | $1.4\times10^{-6}$ | | Gharagheizi et al. (2010) | Q | 246 |
| AFTPEBDOGXRMNQ-UHFFFAOYSA-N | $1.1\times10^{-6}$ | | Hilal et al. (2008) | Q | |
| | $1.6\times10^{-6}$ | | Modarresi et al. (2005) | Q | 247 |
| | $1.0\times10^{-6}$ | | Yao et al. (2002) | Q | 229 |
| | $1.7\times10^{-6}$ | | Yaws (1999) | ? | 21 |
| | $2.1\times10^{-6}$ | | Yaws and Yang (1992) | ? | 21 |
| 2,2,5-trimethylhexane | $1.9\times10^{-6}$ | 6200 | Brockbank (2013) | L | 1 |
| C$_9$H$_{20}$ | $2.8\times10^{-6}$ | | Plyasunov and Shock (2000) | L | |
| [3522-94-9] | $2.9\times10^{-6}$ | | Mackay and Shiu (1981) | L | |
| HHOSMYBYIHNXNO-UHFFFAOYSA-N | $4.1\times10^{-6}$ | | Mackay et al. (2006a) | V | |
| | $4.1\times10^{-6}$ | | Mackay et al. (1993) | V | |
| | $4.1\times10^{-6}$ | | Cabani et al. (1981) | V | |
| | $3.9\times10^{-6}$ | | McAuliffe (1966) | V | |
| | $1.8\times10^{-6}$ | | Yaws (2003) | X | 237 |
| | $1.4\times10^{-6}$ | | Gharagheizi et al. (2012) | Q | |
| | $1.4\times10^{-6}$ | | Gharagheizi et al. (2010) | Q | 246 |
| | $9.0\times10^{-7}$ | | Hilal et al. (2008) | Q | |
| | $1.7\times10^{-6}$ | | Modarresi et al. (2005) | Q | 247 |
| | $2.9\times10^{-6}$ | | Yaffe et al. (2003) | Q | 248, 249 |
| | $1.5\times10^{-6}$ | | Yao et al. (2002) | Q | 229 |
| | $2.2\times10^{-6}$ | | Nirmalakhandan et al. (1997) | Q | |
| | $1.8\times10^{-6}$ | | Yaws (1999) | ? | 21 |
| | $1.9\times10^{-6}$ | | Yaws and Yang (1992) | ? | 21 |
| 2,3,3-trimethylhexane | $1.4\times10^{-6}$ | | Yaws (2003) | X | 237 |
| C$_9$H$_{20}$ | $2.4\times10^{-6}$ | | Gharagheizi et al. (2012) | Q | |
| [16747-28-7] | $1.4\times10^{-6}$ | | Gharagheizi et al. (2010) | Q | 246 |
| DJYSEQMMCZAKGT-UHFFFAOYSA-N | $2.4\times10^{-6}$ | | Hilal et al. (2008) | Q | |
| | $2.3\times10^{-6}$ | | Modarresi et al. (2005) | Q | 247 |
| | $1.0\times10^{-6}$ | | Yao et al. (2002) | Q | 229 |
| | $1.4\times10^{-6}$ | | Yaws (1999) | ? | 21 |
| | $1.7\times10^{-6}$ | | Yaws and Yang (1992) | ? | 21 |
| 2,3,4-trimethylhexane | $1.4\times10^{-6}$ | | Yaws (2003) | X | 237 |
| C$_9$H$_{20}$ | $2.5\times10^{-6}$ | | Gharagheizi et al. (2012) | Q | |
| [921-47-1] | $1.5\times10^{-6}$ | | Gharagheizi et al. (2010) | Q | 246 |
| RUTNOQHQISEBGT-UHFFFAOYSA-N | $2.6\times10^{-6}$ | | Hilal et al. (2008) | Q | |
| | $1.6\times10^{-6}$ | | Modarresi et al. (2005) | Q | 247 |
| | $1.1\times10^{-6}$ | | Yao et al. (2002) | Q | 229 |
| | $1.4\times10^{-6}$ | | Yaws (1999) | ? | 21 |
| | $1.8\times10^{-6}$ | | Yaws and Yang (1992) | ? | 21 |



Table A2.1: Alkanes (...continued)

| Substance / Formula / (Trivial Name) / [CAS Registry Number] / InChIKey | $H_s^{cp}$ (at $T^\ominus$) $\left[\dfrac{\text{mol}}{\text{m}^3\,\text{Pa}}\right]$ | $\dfrac{\text{d}\ln H_s^{cp}}{\text{d}(1/T)}$ [K] | Reference | Type | Note |
|---|---|---|---|---|---|
| 2,3,5-trimethylhexane | $1.7\times10^{-6}$ | | Yaws (2003) | X | 237 |
| $C_9H_{20}$ | $1.8\times10^{-6}$ | | Gharagheizi et al. (2012) | Q | |
| [1069-53-0] | $1.5\times10^{-6}$ | | Gharagheizi et al. (2010) | Q | 246 |
| ODGLTLJZCVNPBU-UHFFFAOYSA-N | $1.4\times10^{-6}$ | | Hilal et al. (2008) | Q | |
| | $2.1\times10^{-6}$ | | Modarresi et al. (2005) | Q | 247 |
| | $1.1\times10^{-6}$ | | Yao et al. (2002) | Q | 229, 267 |
| | $2.8\times10^{-6}$ | | English and Carroll (2001) | Q | 230, 231 |
| | $1.7\times10^{-6}$ | | Yaws (1999) | ? | 21 |
| | $2.0\times10^{-6}$ | | Yaws and Yang (1992) | ? | 21 |
| 2,4,4-trimethylhexane | $1.5\times10^{-6}$ | | Yaws (2003) | X | 237 |
| $C_9H_{20}$ | $1.8\times10^{-6}$ | | Gharagheizi et al. (2012) | Q | |
| [16747-30-1] | $1.4\times10^{-6}$ | | Gharagheizi et al. (2010) | Q | 246 |
| SVEMKBCPZYWEPH-UHFFFAOYSA-N | $1.4\times10^{-6}$ | | Hilal et al. (2008) | Q | |
| | $1.5\times10^{-6}$ | | Modarresi et al. (2005) | Q | 247 |
| | $1.1\times10^{-6}$ | | Yao et al. (2002) | Q | 229 |
| | $1.5\times10^{-6}$ | | Yaws (1999) | ? | 21 |
| | $1.9\times10^{-6}$ | | Yaws and Yang (1992) | ? | 21 |
| 3,3,4-trimethylhexane | $1.3\times10^{-6}$ | | Yaws (2003) | X | 237 |
| $C_9H_{20}$ | $2.7\times10^{-6}$ | | Gharagheizi et al. (2012) | Q | |
| [16747-31-2] | $1.4\times10^{-6}$ | | Gharagheizi et al. (2010) | Q | 246 |
| ARWOOWBJJKVYOV-UHFFFAOYSA-N | $2.9\times10^{-6}$ | | Hilal et al. (2008) | Q | |
| | $1.6\times10^{-6}$ | | Modarresi et al. (2005) | Q | 247 |
| | $1.1\times10^{-6}$ | | Yao et al. (2002) | Q | 229 |
| | $1.3\times10^{-6}$ | | Yaws (1999) | ? | 21 |
| | $1.7\times10^{-6}$ | | Yaws and Yang (1992) | ? | 21 |
| 3-ethyl-2-methylhexane | $1.5\times10^{-6}$ | | Yaws (2003) | X | 237 |
| $C_9H_{20}$ | $2.3\times10^{-6}$ | | Gharagheizi et al. (2012) | Q | |
| [16789-46-1] | $1.5\times10^{-6}$ | | Gharagheizi et al. (2010) | Q | 246 |
| MVLOWDRGPHBNNF-UHFFFAOYSA-N | $2.3\times10^{-6}$ | | Hilal et al. (2008) | Q | |
| | $2.2\times10^{-6}$ | | Modarresi et al. (2005) | Q | 247 |
| | $1.2\times10^{-6}$ | | Yao et al. (2002) | Q | 229 |
| | $1.5\times10^{-6}$ | | Yaws (1999) | ? | 21 |
| | $1.9\times10^{-6}$ | | Yaws and Yang (1992) | ? | 21 |
| 4-ethyl-2-methylhexane | $1.6\times10^{-6}$ | | Yaws (2003) | X | 237 |
| $C_9H_{20}$ | $2.0\times10^{-6}$ | | Gharagheizi et al. (2012) | Q | |
| [3074-75-7] | $1.5\times10^{-6}$ | | Gharagheizi et al. (2010) | Q | 246 |
| KYCZJIBOPKRSOV-UHFFFAOYSA-N | $1.5\times10^{-6}$ | | Hilal et al. (2008) | Q | |
| | $1.6\times10^{-6}$ | | Modarresi et al. (2005) | Q | 247 |
| | $1.5\times10^{-6}$ | | Yao et al. (2002) | Q | 229, 267 |
| | $1.7\times10^{-6}$ | | Yaws (1999) | ? | 21 |
| | $2.0\times10^{-6}$ | | Yaws and Yang (1992) | ? | 21 |





Table A2.1: Alkanes (...continued)

| Substance Formula (Trivial Name) [CAS Registry Number] InChIKey | $H_s^{cp}$ (at $T^\ominus$) $\left[\dfrac{\mathrm{mol}}{\mathrm{m}^3\,\mathrm{Pa}}\right]$ | $\dfrac{\mathrm{d}\ln H_s^{cp}}{\mathrm{d}(1/T)}$ [K] | Reference | Type | Note |
|---|---|---|---|---|---|
| 3-ethyl-3-methylhexane C$_9$H$_{20}$ [3074-76-8] CYWROHZCELEGSE-UHFFFAOYSA-N | $1.3\times10^{-6}$ $2.6\times10^{-6}$ $1.4\times10^{-6}$ $3.2\times10^{-6}$ $1.5\times10^{-6}$ $1.2\times10^{-6}$ $1.3\times10^{-6}$ $1.7\times10^{-6}$ | | Yaws (2003) Gharagheizi et al. (2012) Gharagheizi et al. (2010) Hilal et al. (2008) Modarresi et al. (2005) Yao et al. (2002) Yaws (1999) Yaws and Yang (1992) | X Q Q Q Q Q ? ? | 237 246 247 229 21 21 |
| 3-ethyl-4-methylhexane C$_9$H$_{20}$ [3074-77-9] OKCRKWVABWILDR-UHFFFAOYSA-N | $1.4\times10^{-6}$ $2.6\times10^{-6}$ $1.5\times10^{-6}$ $3.1\times10^{-6}$ $1.9\times10^{-6}$ $1.2\times10^{-6}$ $1.4\times10^{-6}$ $1.8\times10^{-6}$ | | Yaws (2003) Gharagheizi et al. (2012) Gharagheizi et al. (2010) Hilal et al. (2008) Modarresi et al. (2005) Yao et al. (2002) Yaws (1999) Yaws and Yang (1992) | X Q Q Q Q Q ? ? | 237 246 247 229 21 21 |
| 2,2,3,3-tetramethylpentane C$_9$H$_{20}$ [7154-79-2] QUKOJKFJIHSBKV-UHFFFAOYSA-N | $1.2\times10^{-6}$ $2.9\times10^{-6}$ $1.3\times10^{-6}$ $3.6\times10^{-6}$ $2.1\times10^{-6}$ $1.0\times10^{-6}$ $1.2\times10^{-6}$ $1.6\times10^{-6}$ | | Yaws (2003) Gharagheizi et al. (2012) Gharagheizi et al. (2010) Hilal et al. (2008) Modarresi et al. (2005) Yao et al. (2002) Yaws (1999) Yaws and Yang (1992) | X Q Q Q Q Q ? ? | 237 246 247 229, 267 21 21 |
| 2,2,3,4-tetramethylpentane C$_9$H$_{20}$ [1186-53-4] VZFMYOCAEQDWDY-UHFFFAOYSA-N | $1.4\times10^{-6}$ $2.1\times10^{-6}$ $1.5\times10^{-6}$ $1.9\times10^{-6}$ $1.6\times10^{-6}$ $2.8\times10^{-6}$ $1.4\times10^{-6}$ $1.7\times10^{-6}$ | | Yaws (2003) Gharagheizi et al. (2012) Gharagheizi et al. (2010) Hilal et al. (2008) Modarresi et al. (2005) Yao et al. (2002) Yaws (1999) Yaws and Yang (1992) | X Q Q Q Q Q ? ? | 237 246 247 229 21 21 |
| 2,2,4,4-tetramethylpentane C$_9$H$_{20}$ [1070-87-7] GUMULFRCHLJNDY-UHFFFAOYSA-N | $1.7\times10^{-6}$ $1.4\times10^{-6}$ $1.3\times10^{-6}$ $9.0\times10^{-7}$ $1.6\times10^{-6}$ $1.4\times10^{-6}$ $1.7\times10^{-6}$ $1.9\times10^{-6}$ | | Yaws (2003) Gharagheizi et al. (2012) Gharagheizi et al. (2010) Hilal et al. (2008) Modarresi et al. (2005) Yao et al. (2002) Yaws (1999) Yaws and Yang (1992) | X Q Q Q Q Q ? ? | 237 246 247 229 21 21 |



Table A2.1: Alkanes (...continued)

| Substance Formula (Trivial Name) [CAS Registry Number] InChIKey | $H_s^{cp}$ (at $T^{\ominus}$) $\left[\dfrac{\text{mol}}{\text{m}^3\,\text{Pa}}\right]$ | $\dfrac{\text{d}\ln H_s^{cp}}{\text{d}(1/T)}$ [K] | Reference | Type | Note |
|---|---|---|---|---|---|
| 2,3,3,4-tetramethylpentane | $1.2\times10^{-6}$ | | Yaws (2003) | X | 237 |
| $C_9H_{20}$ | $3.0\times10^{-6}$ | | Gharagheizi et al. (2012) | Q | |
| [16747-38-9] | $1.5\times10^{-6}$ | | Gharagheizi et al. (2010) | Q | 246 |
| JLCYYQOQSAMWTA-UHFFFAOYSA-N | $2.7\times10^{-6}$ | | Hilal et al. (2008) | Q | |
| | $1.6\times10^{-6}$ | | Modarresi et al. (2005) | Q | 247 |
| | $1.2\times10^{-6}$ | | Yao et al. (2002) | Q | 229 |
| | $1.2\times10^{-6}$ | | Yaws (1999) | ? | 21 |
| | $1.6\times10^{-6}$ | | Yaws and Yang (1992) | ? | 21 |
| 3-ethyl-2,2-dimethylpentane | $1.5\times10^{-6}$ | | Yaws (2003) | X | 237 |
| $C_9H_{20}$ | $2.2\times10^{-6}$ | | Gharagheizi et al. (2012) | Q | |
| [16747-32-3] | $1.4\times10^{-6}$ | | Gharagheizi et al. (2010) | Q | 246 |
| CLZCPQKGOAXOJT-UHFFFAOYSA-N | $1.9\times10^{-6}$ | | Hilal et al. (2008) | Q | |
| | $1.6\times10^{-6}$ | | Modarresi et al. (2005) | Q | 247 |
| | $1.6\times10^{-6}$ | | Yao et al. (2002) | Q | 229, 267 |
| | $1.5\times10^{-6}$ | | Yaws (1999) | ? | 21 |
| | $1.8\times10^{-6}$ | | Yaws and Yang (1992) | ? | 21 |
| 3-ethyl-2,3-dimethylpentane | $1.2\times10^{-6}$ | | Yaws (2003) | X | 237 |
| $C_9H_{20}$ | $3.2\times10^{-6}$ | | Gharagheizi et al. (2012) | Q | |
| [16747-33-4] | $1.4\times10^{-6}$ | | Gharagheizi et al. (2010) | Q | 246 |
| MMASVVOQIKCFJZ-UHFFFAOYSA-N | $3.5\times10^{-6}$ | | Hilal et al. (2008) | Q | |
| | $1.6\times10^{-6}$ | | Modarresi et al. (2005) | Q | 247 |
| | $1.6\times10^{-6}$ | | Yao et al. (2002) | Q | 229 |
| | $1.2\times10^{-6}$ | | Yaws (1999) | ? | 21 |
| | $1.5\times10^{-6}$ | | Yaws and Yang (1992) | ? | 21 |
| 3-ethyl-2,4-dimethylpentane | $1.4\times10^{-6}$ | | Yaws (2003) | X | 237 |
| $C_9H_{20}$ | $2.3\times10^{-6}$ | | Gharagheizi et al. (2012) | Q | |
| [1068-87-7] | $1.5\times10^{-6}$ | | Gharagheizi et al. (2010) | Q | 246 |
| VLHAGZNBWKUMRW-UHFFFAOYSA-N | $1.9\times10^{-6}$ | | Hilal et al. (2008) | Q | |
| | $2.0\times10^{-6}$ | | Modarresi et al. (2005) | Q | 247 |
| | $1.8\times10^{-6}$ | | Yao et al. (2002) | Q | 229 |
| | $1.4\times10^{-6}$ | | Yaws (1999) | ? | 21 |
| | $1.8\times10^{-6}$ | | Yaws and Yang (1992) | ? | 21 |
| 3,3-diethylpentane | | 4900 | Abraham and Nasehzadeh (1981) | R | |
| $C_9H_{20}$ | $1.1\times10^{-6}$ | | Yaws (2003) | X | 237 |
| [1067-20-5] | $3.4\times10^{-6}$ | | Gharagheizi et al. (2012) | Q | |
| BGXXXYLRPIRDHJ-UHFFFAOYSA-N | $1.4\times10^{-6}$ | | Gharagheizi et al. (2010) | Q | 246 |
| | $4.1\times10^{-6}$ | | Hilal et al. (2008) | Q | |
| | $2.0\times10^{-6}$ | | Modarresi et al. (2005) | Q | 247 |
| | $1.7\times10^{-6}$ | | Yao et al. (2002) | Q | 229 |
| | $1.1\times10^{-6}$ | | Yaws (1999) | ? | 21 |
| | $1.5\times10^{-6}$ | | Yaws and Yang (1992) | ? | 21 |
| | $9.5\times10^{-6}$ | | Abraham et al. (1990) | ? | |
| | $9.4\times10^{-6}$ | | Abraham (1979) | ? | |



Table A2.1: Alkanes (. . . continued)

| Substance Formula (Trivial Name) [CAS Registry Number] InChIKey | $H_s^{cp}$ (at $T^{\ominus}$) $\left[\dfrac{\text{mol}}{\text{m}^3\,\text{Pa}}\right]$ | $\dfrac{\text{d}\ln H_s^{cp}}{\text{d}(1/T)}$ [K] | Reference | Type | Note |
|---|---|---|---|---|---|
| decane | $1.3\times10^{-6}$ | | Brockbank (2013) | L | |
| $C_{10}H_{22}$ | $1.1\times10^{-6}$ | | Plyasunov and Shock (2000) | L | |
| [124-18-5] | $1.4\times10^{-6}$ | | Mackay and Shiu (1981) | L | |
| DIOQZVSQGTUSAI-UHFFFAOYSA-N | $1.9\times10^{-6}$ | | Duchowicz et al. (2020) | V | 186 |
| | $1.9\times10^{-6}$ | | HSDB (2015) | V | |
| | $2.1\times10^{-6}$ | | Mackay et al. (2006a) | V | |
| | $2.1\times10^{-6}$ | | Mackay et al. (1993) | V | |
| | $2.0\times10^{-6}$ | | Hwang et al. (1992) | V | |
| | $2.3\times10^{-6}$ | | Eastcott et al. (1988) | V | |
| | $1.9\times10^{-6}$ | | Abraham (1984) | V | |
| | $1.9\times10^{-6}$ | | Yaws (2003) | X | 258 |
| | $1.9\times10^{-6}$ | | Yaws (2003) | X | 237 |
| | $4.4\times10^{-6}$ | | Dupeux et al. (2022) | Q | 259 |
| | $4.4\times10^{-4}$ | | Duchowicz et al. (2020) | Q | |
| | $2.3\times10^{-5}$ | | Wang et al. (2017) | Q | 80, 238 |
| | $1.7\times10^{-6}$ | | Wang et al. (2017) | Q | 80, 239 |
| | $6.5\times10^{-6}$ | | Wang et al. (2017) | Q | 80, 240 |
| | $3.7\times10^{-6}$ | | Gharagheizi et al. (2012) | Q | |
| | $2.0\times10^{-6}$ | | Raventos-Duran et al. (2010) | Q | 242, 243 |
| | $3.1\times10^{-6}$ | | Raventos-Duran et al. (2010) | Q | 244 |
| | $2.0\times10^{-6}$ | | Raventos-Duran et al. (2010) | Q | 245 |
| | $1.5\times10^{-6}$ | | Gharagheizi et al. (2010) | Q | 246 |
| | $2.2\times10^{-6}$ | | Hilal et al. (2008) | Q | |
| | $3.8\times10^{-6}$ | | Modarresi et al. (2007) | Q | 67 |
| | $2.0\times10^{-6}$ | | Yaffe et al. (2003) | Q | 248, 249 |
| | $2.2\times10^{-6}$ | | English and Carroll (2001) | Q | 230, 231 |
| | $1.0\times10^{-5}$ | | Katritzky et al. (1998) | Q | |
| | $2.9\times10^{-6}$ | | Nirmalakhandan et al. (1997) | Q | |
| | $1.9\times10^{-6}$ | | Yaws (1999) | ? | 21 |
| | $2.1\times10^{-6}$ | | Yaws and Yang (1992) | ? | 21 |
| 2-methylnonane | $1.2\times10^{-6}$ | | Yaws (2003) | X | 237 |
| $C_{10}H_{22}$ | $2.0\times10^{-6}$ | | Gharagheizi et al. (2012) | Q | |
| [871-83-0] | $1.3\times10^{-6}$ | | Gharagheizi et al. (2010) | Q | 246 |
| SGVYKUFIHHTIFL-UHFFFAOYSA-N | $1.5\times10^{-6}$ | | Hilal et al. (2008) | Q | |
| | $1.3\times10^{-6}$ | | Modarresi et al. (2005) | Q | 247 |
| | $3.1\times10^{-6}$ | | Yao et al. (2002) | Q | 229, 267 |
| | $1.2\times10^{-6}$ | | Yaws (1999) | ? | 21 |
| | $1.7\times10^{-6}$ | | Yaws and Yang (1992) | ? | 21 |
| 3-methylnonane | $1.1\times10^{-6}$ | | Yaws (2003) | X | 237 |
| $C_{10}H_{22}$ | $2.1\times10^{-6}$ | | Gharagheizi et al. (2012) | Q | |
| [5911-04-6] | $1.3\times10^{-6}$ | | Gharagheizi et al. (2010) | Q | 246 |
| PLZDDPSCZHRBOY-UHFFFAOYSA-N | $1.7\times10^{-6}$ | | Hilal et al. (2008) | Q | |
| | $1.3\times10^{-6}$ | | Modarresi et al. (2005) | Q | 247 |
| | $2.8\times10^{-6}$ | | Yao et al. (2002) | Q | 229 |
| | $1.1\times10^{-6}$ | | Yaws (1999) | ? | 21 |
| | $1.7\times10^{-6}$ | | Yaws and Yang (1992) | ? | 21 |



Table A2.1: Alkanes (...continued)

| Substance<br>Formula<br>(Trivial Name)<br>[CAS Registry Number]<br>InChIKey | $H_s^{cp}$<br>(at $T^{\ominus}$)<br>$\left[\dfrac{\mathrm{mol}}{\mathrm{m^3\,Pa}}\right]$ | $\dfrac{\mathrm{d}\ln H_s^{cp}}{\mathrm{d}(1/T)}$<br><br>[K] | Reference | Type | Note |
|---|---|---|---|---|---|
| 4-methylnonane | $1.0\times10^{-6}$ | | Yaws (2003) | X | 237 |
| $C_{10}H_{22}$ | $2.0\times10^{-6}$ | | Gharagheizi et al. (2012) | Q | |
| [17301-94-9] | $1.3\times10^{-6}$ | | Gharagheizi et al. (2010) | Q | 246 |
| IALRSQMWHFKJJA-UHFFFAOYSA-N | $1.6\times10^{-6}$ | | Hilal et al. (2008) | Q | |
| | $1.3\times10^{-6}$ | | Modarresi et al. (2005) | Q | 247 |
| | $2.7\times10^{-6}$ | | Yao et al. (2002) | Q | 229 |
| | $1.0\times10^{-6}$ | | Yaws (1999) | ? | 21 |
| | $1.6\times10^{-6}$ | | Yaws and Yang (1992) | ? | 21 |
| 5-methylnonane | $1.1\times10^{-6}$ | | Yaws (2003) | X | 237 |
| $C_{10}H_{22}$ | $1.9\times10^{-6}$ | | Gharagheizi et al. (2012) | Q | |
| [15869-85-9] | $1.3\times10^{-6}$ | | Gharagheizi et al. (2010) | Q | 246 |
| TYSIILFJZXHVPU-UHFFFAOYSA-N | $1.7\times10^{-6}$ | | Hilal et al. (2008) | Q | |
| | $1.3\times10^{-6}$ | | Modarresi et al. (2005) | Q | 247 |
| | $4.2\times10^{-6}$ | | Yao et al. (2002) | Q | 229 |
| | $1.1\times10^{-6}$ | | Yaws (1999) | ? | 21 |
| | $1.6\times10^{-6}$ | | Yaws and Yang (1992) | ? | 21 |
| 2,2-dimethyloctane | $1.1\times10^{-6}$ | | Yaws (2003) | X | 237 |
| $C_{10}H_{22}$ | $1.5\times10^{-6}$ | | Gharagheizi et al. (2012) | Q | |
| [15869-87-1] | $1.1\times10^{-6}$ | | Gharagheizi et al. (2010) | Q | 246 |
| GPBUTTSWJNPYJL-UHFFFAOYSA-N | $1.3\times10^{-6}$ | | Hilal et al. (2008) | Q | |
| | $1.1\times10^{-6}$ | | Yaws (1999) | ? | 21 |
| | $1.7\times10^{-6}$ | | Yaws and Yang (1992) | ? | 21 |
| 2,3-dimethyloctane | $1.0\times10^{-6}$ | | Yaws (2003) | X | 237 |
| $C_{10}H_{22}$ | $1.9\times10^{-6}$ | | Gharagheizi et al. (2012) | Q | |
| [7146-60-3] | $1.2\times10^{-6}$ | | Gharagheizi et al. (2010) | Q | 246 |
| YPMNDMUOGQJCLW-UHFFFAOYSA-N | $1.7\times10^{-6}$ | | Hilal et al. (2008) | Q | |
| | $1.4\times10^{-6}$ | | Modarresi et al. (2005) | Q | 247 |
| | $1.6\times10^{-6}$ | | Yao et al. (2002) | Q | 229 |
| | $1.0\times10^{-6}$ | | Yaws (1999) | ? | 21 |
| | $1.5\times10^{-6}$ | | Yaws and Yang (1992) | ? | 21 |
| 2,4-dimethyloctane | $1.2\times10^{-6}$ | | Yaws (2003) | X | 237 |
| $C_{10}H_{22}$ | $1.5\times10^{-6}$ | | Gharagheizi et al. (2012) | Q | |
| [4032-94-4] | $1.2\times10^{-6}$ | | Gharagheizi et al. (2010) | Q | 246 |
| IXAVTTRPEXFVSX-UHFFFAOYSA-N | $1.2\times10^{-6}$ | | Hilal et al. (2008) | Q | |
| | $1.3\times10^{-6}$ | | Modarresi et al. (2005) | Q | 247 |
| | $1.7\times10^{-6}$ | | Yao et al. (2002) | Q | 229 |
| | $1.2\times10^{-6}$ | | Yaws (1999) | ? | 21 |
| | $1.7\times10^{-6}$ | | Yaws and Yang (1992) | ? | 21 |





Table A2.1: Alkanes (. . . continued)

| Substance<br>Formula<br>(Trivial Name)<br>[CAS Registry Number]<br>InChIKey | $H_s^{cp}$<br>(at $T^\ominus$)<br>$\left[\dfrac{\mathrm{mol}}{\mathrm{m^3\,Pa}}\right]$ | $\dfrac{\mathrm{d}\ln H_s^{cp}}{\mathrm{d}(1/T)}$<br><br>[K] | Reference | Type | Note |
|---|---|---|---|---|---|
| 2,5-dimethyloctane<br>$C_{10}H_{22}$<br>[15869-89-3]<br>HOAAQUNESXYFDT-UHFFFAOYSA-N | $1.1\times10^{-6}$<br>$1.6\times10^{-6}$<br>$1.2\times10^{-6}$<br>$1.3\times10^{-6}$<br>$1.3\times10^{-6}$<br>$2.0\times10^{-6}$<br>$1.1\times10^{-6}$<br>$1.6\times10^{-6}$ | | Yaws (2003)<br>Gharagheizi et al. (2012)<br>Gharagheizi et al. (2010)<br>Hilal et al. (2008)<br>Modarresi et al. (2005)<br>Yao et al. (2002)<br>Yaws (1999)<br>Yaws and Yang (1992) | X<br>Q<br>Q<br>Q<br>Q<br>Q<br>?<br>? | 237<br><br>246<br><br>247<br>229, 267<br>21<br>21 |
| 2,6-dimethyloctane<br>$C_{10}H_{22}$<br>[2051-30-1]<br>ZALHPSXXQIPKTQ-UHFFFAOYSA-N | $1.0\times10^{-6}$<br>$1.7\times10^{-6}$<br>$1.2\times10^{-6}$<br>$1.2\times10^{-6}$<br>$1.3\times10^{-6}$<br>$1.8\times10^{-6}$<br>$1.1\times10^{-6}$<br>$1.6\times10^{-6}$ | | Yaws (2003)<br>Gharagheizi et al. (2012)<br>Gharagheizi et al. (2010)<br>Hilal et al. (2008)<br>Modarresi et al. (2005)<br>Yao et al. (2002)<br>Yaws (1999)<br>Yaws and Yang (1992) | X<br>Q<br>Q<br>Q<br>Q<br>Q<br>?<br>? | 237<br><br>246<br><br>247<br>229<br>21<br>21 |
| 2,7-dimethyloctane<br>$C_{10}H_{22}$<br>[1072-16-8]<br>KEVMYFLMMDUPJE-UHFFFAOYSA-N | $1.2\times10^{-6}$<br>$1.6\times10^{-6}$<br>$1.2\times10^{-6}$<br>$1.0\times10^{-6}$<br>$1.4\times10^{-6}$<br>$1.9\times10^{-6}$<br>$1.2\times10^{-6}$<br>$1.7\times10^{-6}$ | | Yaws (2003)<br>Gharagheizi et al. (2012)<br>Gharagheizi et al. (2010)<br>Hilal et al. (2008)<br>Modarresi et al. (2005)<br>Yao et al. (2002)<br>Yaws (1999)<br>Yaws and Yang (1992) | X<br>Q<br>Q<br>Q<br>Q<br>Q<br>?<br>? | 237<br><br>246<br><br>247<br>229<br>21<br>21 |
| 3,3-dimethyloctane<br>$C_{10}H_{22}$<br>[4110-44-5]<br>DBULLUBYDONGLT-UHFFFAOYSA-N | $1.0\times10^{-6}$<br>$1.8\times10^{-6}$<br>$1.1\times10^{-6}$<br>$1.7\times10^{-6}$<br>$1.3\times10^{-6}$<br>$1.4\times10^{-6}$<br>$1.0\times10^{-6}$<br>$1.5\times10^{-6}$ | | Yaws (2003)<br>Gharagheizi et al. (2012)<br>Gharagheizi et al. (2010)<br>Hilal et al. (2008)<br>Modarresi et al. (2005)<br>Yao et al. (2002)<br>Yaws (1999)<br>Yaws and Yang (1992) | X<br>Q<br>Q<br>Q<br>Q<br>Q<br>?<br>? | 237<br><br>246<br><br>247<br>229<br>21<br>21 |
| 3,4-dimethyloctane<br>$C_{10}H_{22}$<br>[15869-92-8]<br>QQCWGAMGBCGAQJ-UHFFFAOYSA-N | $1.0\times10^{-6}$<br>$1.9\times10^{-6}$<br>$1.2\times10^{-6}$<br>$2.0\times10^{-6}$<br>$1.4\times10^{-6}$<br>$1.6\times10^{-6}$<br>$1.0\times10^{-6}$<br>$1.5\times10^{-6}$ | | Yaws (2003)<br>Gharagheizi et al. (2012)<br>Gharagheizi et al. (2010)<br>Hilal et al. (2008)<br>Modarresi et al. (2005)<br>Yao et al. (2002)<br>Yaws (1999)<br>Yaws and Yang (1992) | X<br>Q<br>Q<br>Q<br>Q<br>Q<br>?<br>? | 237<br><br>246<br><br>247<br>229<br>21<br>21 |



Table A2.1: Alkanes (...continued)

| Substance Formula (Trivial Name) [CAS Registry Number] InChIKey | $H_s^{cp}$ (at $T^{\ominus}$) $\left[\dfrac{\mathrm{mol}}{\mathrm{m^3\,Pa}}\right]$ | $\dfrac{\mathrm{d}\ln H_s^{cp}}{\mathrm{d}(1/T)}$ [K] | Reference | Type | Note |
|---|---|---|---|---|---|
| 3,5-dimethyloctane | $1.1\times10^{-6}$ | | Yaws (2003) | X | 237 |
| $C_{10}H_{22}$ | $1.7\times10^{-6}$ | | Gharagheizi et al. (2012) | Q | |
| [15869-93-9] | $1.2\times10^{-6}$ | | Gharagheizi et al. (2010) | Q | 246 |
| VRHRGVJOUHJULC-UHFFFAOYSA-N | $1.4\times10^{-6}$ | | Hilal et al. (2008) | Q | |
| | $1.4\times10^{-6}$ | | Modarresi et al. (2005) | Q | 247 |
| | $1.6\times10^{-6}$ | | Yao et al. (2002) | Q | 229 |
| | $1.1\times10^{-6}$ | | Yaws (1999) | ? | 21 |
| | $1.6\times10^{-6}$ | | Yaws and Yang (1992) | ? | 21 |
| 3,6-dimethyloctane | $1.1\times10^{-6}$ | | Yaws (2003) | X | 237 |
| $C_{10}H_{22}$ | $1.7\times10^{-6}$ | | Gharagheizi et al. (2012) | Q | |
| [15869-94-0] | $1.2\times10^{-6}$ | | Gharagheizi et al. (2010) | Q | 246 |
| JEEQUUSFXYRPRK-UHFFFAOYSA-N | $1.3\times10^{-6}$ | | Hilal et al. (2008) | Q | |
| | $1.4\times10^{-6}$ | | Modarresi et al. (2005) | Q | 247 |
| | $1.6\times10^{-6}$ | | Yao et al. (2002) | Q | 229 |
| | $1.1\times10^{-6}$ | | Yaws (1999) | ? | 21 |
| | $1.6\times10^{-6}$ | | Yaws and Yang (1992) | ? | 21 |
| 4,4-dimethyloctane | $1.0\times10^{-6}$ | | Yaws (2003) | X | 237 |
| $C_{10}H_{22}$ | $1.6\times10^{-6}$ | | Gharagheizi et al. (2012) | Q | |
| [15869-95-1] | $1.1\times10^{-6}$ | | Gharagheizi et al. (2010) | Q | 246 |
| ZMEDGZAGMLTROM-UHFFFAOYSA-N | $1.5\times10^{-6}$ | | Hilal et al. (2008) | Q | |
| | $1.3\times10^{-6}$ | | Modarresi et al. (2005) | Q | 247 |
| | $1.4\times10^{-6}$ | | Yao et al. (2002) | Q | 229, 267 |
| | $1.1\times10^{-6}$ | | Yaws (1999) | ? | 21 |
| | $1.5\times10^{-6}$ | | Yaws and Yang (1992) | ? | 21 |
| 4,5-dimethyloctane | $1.0\times10^{-6}$ | | Yaws (2003) | X | 237 |
| $C_{10}H_{22}$ | $1.9\times10^{-6}$ | | Gharagheizi et al. (2012) | Q | |
| [15869-96-2] | $1.2\times10^{-6}$ | | Gharagheizi et al. (2010) | Q | 246 |
| DOYJTLUPPPUSMD-UHFFFAOYSA-N | $2.1\times10^{-6}$ | | Hilal et al. (2008) | Q | |
| | $1.3\times10^{-6}$ | | Modarresi et al. (2005) | Q | 247 |
| | $1.4\times10^{-6}$ | | Yao et al. (2002) | Q | 229 |
| | $1.0\times10^{-6}$ | | Yaws (1999) | ? | 21 |
| | $1.5\times10^{-6}$ | | Yaws and Yang (1992) | ? | 21 |
| 3-ethyloctane | $1.1\times10^{-6}$ | | Yaws (2003) | X | 237 |
| $C_{10}H_{22}$ | $2.0\times10^{-6}$ | | Gharagheizi et al. (2012) | Q | |
| [5881-17-4] | $1.3\times10^{-6}$ | | Gharagheizi et al. (2010) | Q | 246 |
| OEYGTUAKNZFCDJ-UHFFFAOYSA-N | $2.2\times10^{-6}$ | | Hilal et al. (2008) | Q | |
| | $1.6\times10^{-6}$ | | Yaws and Yang (1992) | ? | 21 |
| 4-ethyloctane | $1.1\times10^{-6}$ | | Yaws (2003) | X | 237 |
| $C_{10}H_{22}$ | $1.9\times10^{-6}$ | | Gharagheizi et al. (2012) | Q | |
| [15869-86-0] | $1.3\times10^{-6}$ | | Gharagheizi et al. (2010) | Q | 246 |
| NRJUFUBKIFIKFI-UHFFFAOYSA-N | $2.4\times10^{-6}$ | | Hilal et al. (2008) | Q | |
| | $1.3\times10^{-6}$ | | Modarresi et al. (2005) | Q | 247 |
| | $2.8\times10^{-6}$ | | Yao et al. (2002) | Q | 229 |
| | $1.1\times10^{-6}$ | | Yaws (1999) | ? | 21 |
| | $1.6\times10^{-6}$ | | Yaws and Yang (1992) | ? | 21 |





Table A2.1: Alkanes (...continued)

| Substance Formula (Trivial Name) [CAS Registry Number] InChIKey | $H_s^{cp}$ (at $T^\ominus$) $\left[\dfrac{\mathrm{mol}}{\mathrm{m^3\,Pa}}\right]$ | $\dfrac{\mathrm{d}\ln H_s^{cp}}{\mathrm{d}(1/T)}$ [K] | Reference | Type | Note |
|---|---|---|---|---|---|
| 2,2,3-trimethylheptane | $1.0\times10^{-6}$ | | Yaws (2003) | X | 237 |
| $C_{10}H_{22}$ | $1.7\times10^{-6}$ | | Gharagheizi et al. (2012) | Q | |
| [52896-92-1] | $1.1\times10^{-6}$ | | Gharagheizi et al. (2010) | Q | 246 |
| ACYHSTUWOQNWCX-UHFFFAOYSA-N | $1.6\times10^{-6}$ | | Hilal et al. (2008) | Q | |
| | $1.4\times10^{-6}$ | | Modarresi et al. (2005) | Q | 247 |
| | $9.4\times10^{-7}$ | | Yao et al. (2002) | Q | 229 |
| | $1.0\times10^{-6}$ | | Yaws (1999) | ? | 21 |
| | $1.4\times10^{-6}$ | | Yaws and Yang (1992) | ? | 21 |
| 2,2,4-trimethylheptane | $1.2\times10^{-6}$ | | Yaws (2003) | X | 237 |
| $C_{10}H_{22}$ | $1.2\times10^{-6}$ | | Gharagheizi et al. (2012) | Q | |
| [14720-74-2] | $1.1\times10^{-6}$ | | Gharagheizi et al. (2010) | Q | 246 |
| IIYGOARYARWJBO-UHFFFAOYSA-N | $1.3\times10^{-6}$ | | Modarresi et al. (2005) | Q | 247 |
| | $1.0\times10^{-6}$ | | Yao et al. (2002) | Q | 229 |
| | $1.2\times10^{-6}$ | | Yaws (1999) | ? | 21 |
| | $1.6\times10^{-6}$ | | Yaws and Yang (1992) | ? | 21 |
| 2,2,5-trimethylheptane | $1.2\times10^{-6}$ | | Yaws (2003) | X | 237 |
| $C_{10}H_{22}$ | $1.3\times10^{-6}$ | | Gharagheizi et al. (2012) | Q | |
| [20291-95-6] | $1.1\times10^{-6}$ | | Gharagheizi et al. (2010) | Q | 246 |
| GZJFAWOTMWATOS-UHFFFAOYSA-N | $1.5\times10^{-6}$ | | Modarresi et al. (2005) | Q | 247 |
| | $1.1\times10^{-6}$ | | Yao et al. (2002) | Q | 229, 267 |
| | $1.2\times10^{-6}$ | | Yaws (1999) | ? | 21 |
| | $1.6\times10^{-6}$ | | Yaws and Yang (1992) | ? | 21 |
| 2,2,6-trimethylheptane | $1.2\times10^{-6}$ | | Yaws (2003) | X | 237 |
| $C_{10}H_{22}$ | $1.2\times10^{-6}$ | | Gharagheizi et al. (2012) | Q | |
| [1190-83-6] | $1.1\times10^{-6}$ | | Gharagheizi et al. (2010) | Q | 246 |
| FHJCGIUZJXWNET-UHFFFAOYSA-N | $1.4\times10^{-6}$ | | Modarresi et al. (2005) | Q | 247 |
| | $1.3\times10^{-6}$ | | Yao et al. (2002) | Q | 229 |
| | $1.3\times10^{-6}$ | | Yaws (1999) | ? | 21 |
| | $1.7\times10^{-6}$ | | Yaws and Yang (1992) | ? | 21 |
| 2,3,3-trimethylheptane | $9.5\times10^{-7}$ | | Yaws (2003) | X | 237 |
| $C_{10}H_{22}$ | $1.9\times10^{-6}$ | | Gharagheizi et al. (2012) | Q | |
| [52896-93-2] | $1.1\times10^{-6}$ | | Gharagheizi et al. (2010) | Q | 246 |
| QACXEXNKLFWKLK-UHFFFAOYSA-N | $1.6\times10^{-6}$ | | Hilal et al. (2008) | Q | |
| | $1.3\times10^{-6}$ | | Modarresi et al. (2005) | Q | 247 |
| | $9.7\times10^{-7}$ | | Yao et al. (2002) | Q | 229 |
| | $9.5\times10^{-7}$ | | Yaws (1999) | ? | 21 |
| | $1.4\times10^{-6}$ | | Yaws and Yang (1992) | ? | 21 |
| 2,3,4-trimethylheptane | $9.9\times10^{-7}$ | | Yaws (2003) | X | 237 |
| $C_{10}H_{22}$ | $1.8\times10^{-6}$ | | Gharagheizi et al. (2012) | Q | |
| [52896-95-4] | $1.2\times10^{-6}$ | | Gharagheizi et al. (2010) | Q | 246 |
| UVVYAKOLFKEZEE-UHFFFAOYSA-N | $1.6\times10^{-6}$ | | Hilal et al. (2008) | Q | |
| | $1.4\times10^{-6}$ | | Modarresi et al. (2005) | Q | 247 |
| | $9.0\times10^{-7}$ | | Yao et al. (2002) | Q | 229 |
| | $9.9\times10^{-7}$ | | Yaws (1999) | ? | 21 |



Table A2.1: Alkanes (...continued)

| Substance Formula (Trivial Name) [CAS Registry Number] InChIKey | $H_s^{cp}$ (at $T^{\ominus}$) $\left[\dfrac{\text{mol}}{\text{m}^3\,\text{Pa}}\right]$ | $\dfrac{\text{d}\ln H_s^{cp}}{\text{d}(1/T)}$ [K] | Reference | Type | Note |
|---|---|---|---|---|---|
| | $1.4\times10^{-6}$ | | Yaws and Yang (1992) | ? | 21 |
| 2,3,5-trimethylheptane | $9.5\times10^{-7}$ | | Yaws (2003) | X | 237 |
| $C_{10}H_{22}$ | $1.8\times10^{-6}$ | | Gharagheizi et al. (2012) | Q | |
| [20278-85-7] | $1.2\times10^{-6}$ | | Gharagheizi et al. (2010) | Q | 246 |
| YKPNYFKOKKKGNM-UHFFFAOYSA-N | $1.1\times10^{-6}$ | | Hilal et al. (2008) | Q | |
| | $1.4\times10^{-6}$ | | Modarresi et al. (2005) | Q | 247 |
| | $1.1\times10^{-6}$ | | Yao et al. (2002) | Q | 229 |
| | $9.5\times10^{-7}$ | | Yaws (1999) | ? | 21 |
| | $1.4\times10^{-6}$ | | Yaws and Yang (1992) | ? | 21 |
| 2,3,6-trimethylheptane | $1.1\times10^{-6}$ | | Yaws (2003) | X | 237 |
| $C_{10}H_{22}$ | $1.5\times10^{-6}$ | | Gharagheizi et al. (2012) | Q | |
| [4032-93-3] | $1.2\times10^{-6}$ | | Gharagheizi et al. (2010) | Q | 246 |
| IHPXJGBVRWFEJB-UHFFFAOYSA-N | $1.1\times10^{-6}$ | | Hilal et al. (2008) | Q | |
| | $1.4\times10^{-6}$ | | Modarresi et al. (2005) | Q | 247 |
| | $1.2\times10^{-6}$ | | Yao et al. (2002) | Q | 229 |
| | $1.1\times10^{-6}$ | | Yaws (1999) | ? | 21 |
| | $1.6\times10^{-6}$ | | Yaws and Yang (1992) | ? | 21 |
| 2,4,4-trimethylheptane | $1.1\times10^{-6}$ | | Yaws (2003) | X | 237 |
| $C_{10}H_{22}$ | $1.3\times10^{-6}$ | | Gharagheizi et al. (2012) | Q | |
| [4032-92-2] | $1.1\times10^{-6}$ | | Gharagheizi et al. (2010) | Q | 246 |
| QALGVLROELGEEM-UHFFFAOYSA-N | $1.1\times10^{-6}$ | | Hilal et al. (2008) | Q | |
| | $1.3\times10^{-6}$ | | Modarresi et al. (2005) | Q | 247 |
| | $8.6\times10^{-7}$ | | Yao et al. (2002) | Q | 229 |
| | $1.1\times10^{-6}$ | | Yaws (1999) | ? | 21 |
| | $1.5\times10^{-6}$ | | Yaws and Yang (1992) | ? | 21 |
| 2,4,5-trimethylheptane | $1.1\times10^{-6}$ | | Yaws (2003) | X | 237 |
| $C_{10}H_{22}$ | $1.6\times10^{-6}$ | | Gharagheizi et al. (2012) | Q | |
| [20278-84-6] | $1.2\times10^{-6}$ | | Gharagheizi et al. (2010) | Q | 246 |
| YMBNRMDSLJNNPF-UHFFFAOYSA-N | $1.1\times10^{-6}$ | | Hilal et al. (2008) | Q | |
| | $1.3\times10^{-6}$ | | Modarresi et al. (2005) | Q | 247 |
| | $9.0\times10^{-7}$ | | Yao et al. (2002) | Q | 229, 267 |
| | $1.1\times10^{-6}$ | | Yaws (1999) | ? | 21 |
| | $1.5\times10^{-6}$ | | Yaws and Yang (1992) | ? | 21 |
| 2,4,6-trimethylheptane | $1.3\times10^{-6}$ | | Yaws (2003) | X | 237 |
| $C_{10}H_{22}$ | $1.2\times10^{-6}$ | | Gharagheizi et al. (2012) | Q | |
| [2613-61-8] | $1.2\times10^{-6}$ | | Gharagheizi et al. (2010) | Q | 246 |
| YNLBBDHDNIXQNL-UHFFFAOYSA-N | $7.5\times10^{-7}$ | | Hilal et al. (2008) | Q | |
| | $1.3\times10^{-6}$ | | Modarresi et al. (2005) | Q | 247 |
| | $1.0\times10^{-6}$ | | Yao et al. (2002) | Q | 229 |
| | $1.3\times10^{-6}$ | | Yaws (1999) | ? | 21 |
| | $1.8\times10^{-6}$ | | Yaws and Yang (1992) | ? | 21 |



Table A2.1: Alkanes (. . . continued)

| Substance<br>Formula<br>(Trivial Name)<br>[CAS Registry Number]<br>InChIKey | $H_s^{cp}$<br>(at $T^\ominus$)<br>$\left[\dfrac{\mathrm{mol}}{\mathrm{m}^3\,\mathrm{Pa}}\right]$ | $\dfrac{\mathrm{d}\ln H_s^{cp}}{\mathrm{d}(1/T)}$<br><br>[K] | Reference | Type | Note |
|---|---|---|---|---|---|
| 2,5,5-trimethylheptane<br>$C_{10}H_{22}$<br>[1189-99-7]<br>SOYLPZSOEXZMLE-UHFFFAOYSA-N | $1.1\times10^{-6}$<br>$1.4\times10^{-6}$<br>$1.1\times10^{-6}$<br>$1.1\times10^{-6}$<br>$1.3\times10^{-6}$<br>$1.2\times10^{-6}$<br>$1.1\times10^{-6}$<br>$1.5\times10^{-6}$ | | Yaws (2003)<br>Gharagheizi et al. (2012)<br>Gharagheizi et al. (2010)<br>Hilal et al. (2008)<br>Modarresi et al. (2005)<br>Yao et al. (2002)<br>Yaws (1999)<br>Yaws and Yang (1992) | X<br>Q<br>Q<br>Q<br>Q<br>Q<br>?<br>? | 237<br><br>246<br><br>247<br>229<br>21<br>21 |
| 3,3,4-trimethylheptane<br>$C_{10}H_{22}$<br>[20278-87-9]<br>WRBHKVWLEIYLDZ-UHFFFAOYSA-N | $9.0\times10^{-7}$<br>$2.0\times10^{-6}$<br>$1.1\times10^{-6}$<br>$1.9\times10^{-6}$<br>$1.4\times10^{-6}$<br>$7.6\times10^{-7}$<br>$9.0\times10^{-7}$<br>$1.3\times10^{-6}$ | | Yaws (2003)<br>Gharagheizi et al. (2012)<br>Gharagheizi et al. (2010)<br>Hilal et al. (2008)<br>Modarresi et al. (2005)<br>Yao et al. (2002)<br>Yaws (1999)<br>Yaws and Yang (1992) | X<br>Q<br>Q<br>Q<br>Q<br>Q<br>?<br>? | 237<br><br>246<br><br>247<br>229<br>21<br>21 |
| 3,3,5-trimethylheptane<br>$C_{10}H_{22}$<br>[7154-80-5]<br>VRVRZZWPKABUOE-UHFFFAOYSA-N | $9.9\times10^{-7}$<br>$1.6\times10^{-6}$<br>$1.1\times10^{-6}$<br>$1.2\times10^{-6}$<br>$1.3\times10^{-6}$<br>$8.3\times10^{-7}$<br>$1.0\times10^{-6}$<br>$1.4\times10^{-6}$ | | Yaws (2003)<br>Gharagheizi et al. (2012)<br>Gharagheizi et al. (2010)<br>Hilal et al. (2008)<br>Modarresi et al. (2005)<br>Yao et al. (2002)<br>Yaws (1999)<br>Yaws and Yang (1992) | X<br>Q<br>Q<br>Q<br>Q<br>Q<br>?<br>? | 237<br><br>246<br><br>247<br>229<br>21<br>21 |
| 3,4,4-trimethylheptane<br>$C_{10}H_{22}$<br>[20278-88-0]<br>BLNBSBLKPFFJKQ-UHFFFAOYSA-N | $9.0\times10^{-7}$<br>$2.0\times10^{-6}$<br>$1.1\times10^{-6}$<br>$1.9\times10^{-6}$<br>$1.3\times10^{-6}$<br>$7.7\times10^{-7}$<br>$9.0\times10^{-7}$<br>$1.3\times10^{-6}$ | | Yaws (2003)<br>Gharagheizi et al. (2012)<br>Gharagheizi et al. (2010)<br>Hilal et al. (2008)<br>Modarresi et al. (2005)<br>Yao et al. (2002)<br>Yaws (1999)<br>Yaws and Yang (1992) | X<br>Q<br>Q<br>Q<br>Q<br>Q<br>?<br>? | 237<br><br>246<br><br>247<br>229<br>21<br>21 |
| 3,4,5-trimethylheptane<br>$C_{10}H_{22}$<br>[20278-89-1]<br>LJIIBBYARMPSMT-UHFFFAOYSA-N | $9.4\times10^{-7}$<br>$1.9\times10^{-6}$<br>$1.2\times10^{-6}$<br>$2.3\times10^{-6}$<br>$1.4\times10^{-6}$<br>$8.9\times10^{-7}$<br>$9.4\times10^{-7}$<br>$1.4\times10^{-6}$ | | Yaws (2003)<br>Gharagheizi et al. (2012)<br>Gharagheizi et al. (2010)<br>Hilal et al. (2008)<br>Modarresi et al. (2005)<br>Yao et al. (2002)<br>Yaws (1999)<br>Yaws and Yang (1992) | X<br>Q<br>Q<br>Q<br>Q<br>Q<br>?<br>? | 237<br><br>246<br><br>247<br>229<br>21<br>21 |



Table A2.1: Alkanes (. . . continued)

| Substance<br>Formula<br>(Trivial Name)<br>[CAS Registry Number]<br>InChIKey | $H_\mathrm{s}^{cp}$<br>(at $T^\ominus$)<br>$\left[\dfrac{\mathrm{mol}}{\mathrm{m^3\,Pa}}\right]$ | $\dfrac{\mathrm{d}\ln H_\mathrm{s}^{cp}}{\mathrm{d}(1/T)}$<br><br>[K] | Reference | Type | Note |
|---|---|---|---|---|---|
| 3-ethyl-2-methylheptane | $1.0\times10^{-6}$ | | Yaws (2003) | X | 237 |
| $C_{10}H_{22}$ | $1.8\times10^{-6}$ | | Gharagheizi et al. (2012) | Q | |
| [14676-29-0] | $1.2\times10^{-6}$ | | Gharagheizi et al. (2010) | Q | 246 |
| NKMJCVVUYDKHAV-UHFFFAOYSA-N | $2.0\times10^{-6}$ | | Hilal et al. (2008) | Q | |
| | $1.3\times10^{-6}$ | | Modarresi et al. (2005) | Q | 247 |
| | $1.4\times10^{-6}$ | | Yao et al. (2002) | Q | 229 |
| | $1.0\times10^{-6}$ | | Yaws (1999) | ? | 21 |
| | $1.5\times10^{-6}$ | | Yaws and Yang (1992) | ? | 21 |
| 4-ethyl-2-methylheptane | $1.1\times10^{-6}$ | | Yaws (2003) | X | 237 |
| $C_{10}H_{22}$ | $1.5\times10^{-6}$ | | Gharagheizi et al. (2012) | Q | |
| [52896-88-5] | $1.2\times10^{-6}$ | | Gharagheizi et al. (2010) | Q | 246 |
| OJDKRASKNKPYDH-UHFFFAOYSA-N | $1.4\times10^{-6}$ | | Hilal et al. (2008) | Q | |
| | $1.3\times10^{-6}$ | | Modarresi et al. (2005) | Q | 247 |
| | $1.5\times10^{-6}$ | | Yao et al. (2002) | Q | 229 |
| | $1.1\times10^{-6}$ | | Yaws (1999) | ? | 21 |
| | $1.6\times10^{-6}$ | | Yaws and Yang (1992) | ? | 21 |
| 5-ethyl-2-methylheptane | $1.1\times10^{-6}$ | | Yaws (2003) | X | 237 |
| $C_{10}H_{22}$ | $1.7\times10^{-6}$ | | Gharagheizi et al. (2012) | Q | |
| [13475-78-0] | $1.2\times10^{-6}$ | | Gharagheizi et al. (2010) | Q | 246 |
| DGEMPTLPTFNEHJ-UHFFFAOYSA-N | $1.4\times10^{-6}$ | | Hilal et al. (2008) | Q | |
| | $1.4\times10^{-6}$ | | Modarresi et al. (2005) | Q | 247 |
| | $1.5\times10^{-6}$ | | Yao et al. (2002) | Q | 229 |
| | $1.1\times10^{-6}$ | | Yaws (1999) | ? | 21 |
| | $1.6\times10^{-6}$ | | Yaws and Yang (1992) | ? | 21 |
| 3-ethyl-3-methylheptane | $9.1\times10^{-7}$ | | Yaws (2003) | X | 237 |
| $C_{10}H_{22}$ | $2.0\times10^{-6}$ | | Gharagheizi et al. (2012) | Q | |
| [17302-01-1] | $1.1\times10^{-6}$ | | Gharagheizi et al. (2010) | Q | 246 |
| HSOMNBKXPGCNBH-UHFFFAOYSA-N | $2.2\times10^{-6}$ | | Hilal et al. (2008) | Q | |
| | $1.3\times10^{-6}$ | | Modarresi et al. (2005) | Q | 247 |
| | $1.3\times10^{-6}$ | | Yao et al. (2002) | Q | 229, 267 |
| | $9.2\times10^{-7}$ | | Yaws (1999) | ? | 21 |
| | $1.4\times10^{-6}$ | | Yaws and Yang (1992) | ? | 21 |
| 4-ethyl-3-methylheptane | $9.8\times10^{-7}$ | | Yaws (2003) | X | 237 |
| $C_{10}H_{22}$ | $1.9\times10^{-6}$ | | Gharagheizi et al. (2012) | Q | |
| [52896-89-6] | $1.2\times10^{-6}$ | | Gharagheizi et al. (2010) | Q | 246 |
| BTGGSWBKRYMHQK-UHFFFAOYSA-N | $2.2\times10^{-6}$ | | Hilal et al. (2008) | Q | |
| | $1.4\times10^{-6}$ | | Modarresi et al. (2005) | Q | 247 |
| | $1.2\times10^{-6}$ | | Yao et al. (2002) | Q | 229 |
| | $9.8\times10^{-7}$ | | Yaws (1999) | ? | 21 |
| | $1.4\times10^{-6}$ | | Yaws and Yang (1992) | ? | 21 |



Table A2.1: Alkanes (. . . continued)

| Substance Formula (Trivial Name) [CAS Registry Number] InChIKey | $H_s^{cp}$ (at $T^\ominus$) $\left[\dfrac{\text{mol}}{\text{m}^3\,\text{Pa}}\right]$ | $\dfrac{\text{d}\ln H_s^{cp}}{\text{d}(1/T)}$ [K] | Reference | Type | Note |
|---|---|---|---|---|---|
| 3-ethyl-5-methylheptane | $1.1\times10^{-6}$ | | Yaws (2003) | X | 237 |
| $C_{10}H_{22}$ | $1.7\times10^{-6}$ | | Gharagheizi et al. (2012) | Q | |
| [52896-90-9] | $1.2\times10^{-6}$ | | Gharagheizi et al. (2010) | Q | 246 |
| VXARVYMIZCGZGG-UHFFFAOYSA-N | $1.3\times10^{-6}$ | | Hilal et al. (2008) | Q | |
| | $1.3\times10^{-6}$ | | Modarresi et al. (2005) | Q | 247 |
| | $1.6\times10^{-6}$ | | Yao et al. (2002) | Q | 229 |
| | $1.1\times10^{-6}$ | | Yaws (1999) | ? | 21 |
| | $1.6\times10^{-6}$ | | Yaws and Yang (1992) | ? | 21 |
| 3-ethyl-4-methylheptane | $9.7\times10^{-7}$ | | Yaws (2003) | X | 237 |
| $C_{10}H_{22}$ | $2.0\times10^{-6}$ | | Gharagheizi et al. (2012) | Q | |
| [52896-91-0] | $1.2\times10^{-6}$ | | Gharagheizi et al. (2010) | Q | 246 |
| JZBKRUIGSVOOIC-UHFFFAOYSA-N | $2.2\times10^{-6}$ | | Hilal et al. (2008) | Q | |
| | $1.3\times10^{-6}$ | | Modarresi et al. (2005) | Q | 247 |
| | $1.1\times10^{-6}$ | | Yao et al. (2002) | Q | 229 |
| | $9.8\times10^{-7}$ | | Yaws (1999) | ? | 21 |
| | $1.4\times10^{-6}$ | | Yaws and Yang (1992) | ? | 21 |
| 4-ethyl-4-methylheptane | $9.5\times10^{-7}$ | | Yaws (2003) | X | 237 |
| $C_{10}H_{22}$ | $1.9\times10^{-6}$ | | Gharagheizi et al. (2012) | Q | |
| [17302-04-4] | $1.1\times10^{-6}$ | | Gharagheizi et al. (2010) | Q | 246 |
| MPYQJQDSICRCJJ-UHFFFAOYSA-N | $2.4\times10^{-6}$ | | Hilal et al. (2008) | Q | |
| | $1.3\times10^{-6}$ | | Modarresi et al. (2005) | Q | 247 |
| | $1.0\times10^{-6}$ | | Yao et al. (2002) | Q | 229 |
| | $9.5\times10^{-7}$ | | Yaws (1999) | ? | 21 |
| | $1.4\times10^{-6}$ | | Yaws and Yang (1992) | ? | 21 |
| 4-propylheptane | $1.2\times10^{-6}$ | | Yaws (2003) | X | 237 |
| $C_{10}H_{22}$ | $1.6\times10^{-6}$ | | Gharagheizi et al. (2012) | Q | |
| [3178-29-8] | $1.3\times10^{-6}$ | | Gharagheizi et al. (2010) | Q | 246 |
| ABYGSZMCWVXFCQ-UHFFFAOYSA-N | $1.6\times10^{-6}$ | | Hilal et al. (2008) | Q | |
| | $1.3\times10^{-6}$ | | Modarresi et al. (2005) | Q | 247 |
| | $2.6\times10^{-6}$ | | Yao et al. (2002) | Q | 229 |
| | $1.2\times10^{-6}$ | | Yaws (1999) | ? | 21 |
| | $1.7\times10^{-6}$ | | Yaws and Yang (1992) | ? | 21 |
| 4-(1-methylethyl)-heptane | $1.0\times10^{-6}$ | | Yaws (2003) | X | 237 |
| $C_{10}H_{22}$ | $1.7\times10^{-6}$ | | Gharagheizi et al. (2012) | Q | |
| (4-isopropylheptane) | $1.2\times10^{-6}$ | | Gharagheizi et al. (2010) | Q | 246 |
| [52896-87-4] | $2.1\times10^{-6}$ | | Hilal et al. (2008) | Q | |
| AZLAWGCUDHUQDB-UHFFFAOYSA-N | $1.4\times10^{-6}$ | | Modarresi et al. (2005) | Q | 247 |
| | $2.0\times10^{-6}$ | | Yao et al. (2002) | Q | 229 |
| | $1.0\times10^{-6}$ | | Yaws (1999) | ? | 21 |
| | $1.5\times10^{-6}$ | | Yaws and Yang (1992) | ? | 21 |



Table A2.1: Alkanes (... continued)

| Substance Formula (Trivial Name) [CAS Registry Number] InChIKey | $H_s^{cp}$ (at $T^{\ominus}$) $\left[\dfrac{\mathrm{mol}}{\mathrm{m}^3\,\mathrm{Pa}}\right]$ | $\dfrac{\mathrm{d}\ln H_s^{cp}}{\mathrm{d}(1/T)}$ [K] | Reference | Type | Note |
|---|---|---|---|---|---|
| 2,2,3,3-tetramethylhexane | $8.4\times10^{-7}$ | | Yaws (2003) | X | 237 |
| $C_{10}H_{22}$ | $2.1\times10^{-6}$ | | Gharagheizi et al. (2012) | Q | |
| [13475-81-5] | $9.9\times10^{-7}$ | | Gharagheizi et al. (2010) | Q | 246 |
| RMQHJMMCLSJULX-UHFFFAOYSA-N | $1.8\times10^{-6}$ | | Hilal et al. (2008) | Q | |
| | $1.5\times10^{-6}$ | | Modarresi et al. (2005) | Q | 247 |
| | $4.9\times10^{-7}$ | | Yao et al. (2002) | Q | 229 |
| | $8.4\times10^{-7}$ | | Yaws (1999) | ? | 21 |
| | $1.2\times10^{-6}$ | | Yaws and Yang (1992) | ? | 21 |
| 2,2,3,4-tetramethylhexane | $8.7\times10^{-7}$ | | Yaws (2003) | X | 237 |
| $C_{10}H_{22}$ | $1.8\times10^{-6}$ | | Gharagheizi et al. (2012) | Q | |
| [52897-08-2] | $1.1\times10^{-6}$ | | Gharagheizi et al. (2010) | Q | 246 |
| MHPSPNGWFAGBNH-UHFFFAOYSA-N | $1.2\times10^{-6}$ | | Hilal et al. (2008) | Q | |
| | $1.4\times10^{-6}$ | | Modarresi et al. (2005) | Q | 247 |
| | $5.9\times10^{-7}$ | | Yao et al. (2002) | Q | 229 |
| | $8.7\times10^{-7}$ | | Yaws (1999) | ? | 21 |
| | $1.2\times10^{-6}$ | | Yaws and Yang (1992) | ? | 21 |
| 2,2,3,5-tetramethylhexane | $1.1\times10^{-6}$ | | Yaws (2003) | X | 237 |
| $C_{10}H_{22}$ | $1.3\times10^{-6}$ | | Gharagheizi et al. (2012) | Q | |
| [52897-09-3] | $1.1\times10^{-6}$ | | Gharagheizi et al. (2010) | Q | 246 |
| GCFKTDRTZYDRBI-UHFFFAOYSA-N | $8.4\times10^{-7}$ | | Hilal et al. (2008) | Q | |
| | $1.4\times10^{-6}$ | | Modarresi et al. (2005) | Q | 247 |
| | $6.6\times10^{-7}$ | | Yao et al. (2002) | Q | 229, 267 |
| | $1.1\times10^{-6}$ | | Yaws (1999) | ? | 21 |
| | $1.6\times10^{-6}$ | | Yaws and Yang (1992) | ? | 21 |
| 2,2,4,4-tetramethylhexane | $8.3\times10^{-7}$ | | Yaws (2003) | X | 237 |
| $C_{10}H_{22}$ | $1.5\times10^{-6}$ | | Gharagheizi et al. (2012) | Q | |
| [51750-65-3] | $9.9\times10^{-7}$ | | Gharagheizi et al. (2010) | Q | 246 |
| PXHNHTBJHHSVPT-UHFFFAOYSA-N | $8.8\times10^{-7}$ | | Hilal et al. (2008) | Q | |
| | $1.3\times10^{-6}$ | | Modarresi et al. (2005) | Q | 247 |
| | $6.5\times10^{-7}$ | | Yao et al. (2002) | Q | 229 |
| | $8.3\times10^{-7}$ | | Yaws (1999) | ? | 21 |
| | $1.1\times10^{-6}$ | | Yaws and Yang (1992) | ? | 21 |
| 2,2,4,5-tetramethylhexane | $1.1\times10^{-6}$ | | Yaws (2003) | X | 237 |
| $C_{10}H_{22}$ | $1.2\times10^{-6}$ | | Gharagheizi et al. (2012) | Q | |
| [16747-42-5] | $1.1\times10^{-6}$ | | Gharagheizi et al. (2010) | Q | 246 |
| KDRZICOOQNIJDN-UHFFFAOYSA-N | $8.0\times10^{-7}$ | | Hilal et al. (2008) | Q | |
| | $1.3\times10^{-6}$ | | Modarresi et al. (2005) | Q | 247 |
| | $7.3\times10^{-7}$ | | Yao et al. (2002) | Q | 229 |
| | $1.1\times10^{-6}$ | | Yaws (1999) | ? | 21 |
| | $1.5\times10^{-6}$ | | Yaws and Yang (1992) | ? | 21 |



Table A2.1: Alkanes (. . . continued)

| Substance Formula (Trivial Name) [CAS Registry Number] InChIKey | $H_s^{cp}$ (at $T^{\ominus}$) $\left[\dfrac{\mathrm{mol}}{\mathrm{m}^3\,\mathrm{Pa}}\right]$ | $\dfrac{\mathrm{d}\ln H_s^{cp}}{\mathrm{d}(1/T)}$ [K] | Reference | Type | Note |
|---|---|---|---|---|---|
| 2,2,5,5-tetramethylhexane | $1.4\times10^{-6}$ | | Yaws (2003) | X | 237 |
| $C_{10}H_{22}$ | $8.9\times10^{-7}$ | | Gharagheizi et al. (2012) | Q | |
| [1071-81-4] | $9.9\times10^{-7}$ | | Gharagheizi et al. (2010) | Q | 246 |
| HXQDUXXBVMMIKL-UHFFFAOYSA-N | $4.6\times10^{-7}$ | | Hilal et al. (2008) | Q | |
| | $1.5\times10^{-6}$ | | Modarresi et al. (2005) | Q | 247 |
| | $1.1\times10^{-6}$ | | Yao et al. (2002) | Q | 229 |
| | $1.4\times10^{-6}$ | | Yaws (1999) | ? | 21 |
| | $1.8\times10^{-6}$ | | Yaws and Yang (1992) | ? | 21 |
| 2,3,3,4-tetramethylhexane | $7.9\times10^{-7}$ | | Yaws (2003) | X | 237 |
| $C_{10}H_{22}$ | $2.3\times10^{-6}$ | | Gharagheizi et al. (2012) | Q | |
| [52897-10-6] | $1.1\times10^{-6}$ | | Gharagheizi et al. (2010) | Q | 246 |
| HIHSOGFAVTVMCY-UHFFFAOYSA-N | $1.4\times10^{-6}$ | | Hilal et al. (2008) | Q | |
| | $1.4\times10^{-6}$ | | Modarresi et al. (2005) | Q | 247 |
| | $6.0\times10^{-7}$ | | Yao et al. (2002) | Q | 229 |
| | $7.9\times10^{-7}$ | | Yaws (1999) | ? | 21 |
| | $1.2\times10^{-6}$ | | Yaws and Yang (1992) | ? | 21 |
| 2,3,3,5-tetramethylhexane | $1.0\times10^{-6}$ | | Yaws (2003) | X | 237 |
| $C_{10}H_{22}$ | $1.5\times10^{-6}$ | | Gharagheizi et al. (2012) | Q | |
| [52897-11-7] | $1.1\times10^{-6}$ | | Gharagheizi et al. (2010) | Q | 246 |
| GCGFXFIPOBRMQT-UHFFFAOYSA-N | $9.2\times10^{-7}$ | | Hilal et al. (2008) | Q | |
| | $1.3\times10^{-6}$ | | Modarresi et al. (2005) | Q | 247 |
| | $5.9\times10^{-7}$ | | Yao et al. (2002) | Q | 229 |
| | $1.0\times10^{-6}$ | | Yaws (1999) | ? | 21 |
| | $1.4\times10^{-6}$ | | Yaws and Yang (1992) | ? | 21 |
| 2,3,4,4-tetramethylhexane | $8.2\times10^{-7}$ | | Yaws (2003) | X | 237 |
| $C_{10}H_{22}$ | $2.1\times10^{-6}$ | | Gharagheizi et al. (2012) | Q | |
| [52897-12-8] | $1.1\times10^{-6}$ | | Gharagheizi et al. (2010) | Q | 246 |
| XDRDDPSGUQMOBO-UHFFFAOYSA-N | $1.3\times10^{-6}$ | | Hilal et al. (2008) | Q | |
| | $1.5\times10^{-6}$ | | Modarresi et al. (2005) | Q | 247 |
| | $6.3\times10^{-7}$ | | Yao et al. (2002) | Q | 229 |
| | $8.3\times10^{-7}$ | | Yaws (1999) | ? | 21 |
| | $1.2\times10^{-6}$ | | Yaws and Yang (1992) | ? | 21 |
| 2,3,4,5-tetramethylhexane | $1.0\times10^{-6}$ | | Yaws (2003) | X | 237 |
| $C_{10}H_{22}$ | $1.7\times10^{-6}$ | | Gharagheizi et al. (2012) | Q | |
| [52897-15-1] | $1.2\times10^{-6}$ | | Gharagheizi et al. (2010) | Q | 246 |
| BHGNYYIOYPFWKC-UHFFFAOYSA-N | $1.2\times10^{-6}$ | | Hilal et al. (2008) | Q | |
| | $1.4\times10^{-6}$ | | Modarresi et al. (2005) | Q | 247 |
| | $6.6\times10^{-7}$ | | Yao et al. (2002) | Q | 229, 267 |
| | $1.0\times10^{-6}$ | | Yaws (1999) | ? | 21 |
| | $1.4\times10^{-6}$ | | Yaws and Yang (1992) | ? | 21 |



Table A2.1: Alkanes (...continued)

| Substance Formula (Trivial Name) [CAS Registry Number] InChIKey | $H_s^{cp}$ (at $T^{\ominus}$) $\left[\dfrac{\mathrm{mol}}{\mathrm{m^3\,Pa}}\right]$ | $\dfrac{\mathrm{d}\ln H_s^{cp}}{\mathrm{d}(1/T)}$ [K] | Reference | Type | Note |
|---|---|---|---|---|---|
| 3,3,4,4-tetramethylhexane | $6.6\times10^{-7}$ | | Yaws (2003) | X | 237 |
| C$_{10}$H$_{22}$ | $2.9\times10^{-6}$ | | Gharagheizi et al. (2012) | Q | |
| [5171-84-6] | $9.9\times10^{-7}$ | | Gharagheizi et al. (2010) | Q | 246 |
| MCEYLFHKATVXLN-UHFFFAOYSA-N | $2.2\times10^{-6}$ | | Hilal et al. (2008) | Q | |
| | $1.5\times10^{-6}$ | | Modarresi et al. (2005) | Q | 247 |
| | $6.2\times10^{-7}$ | | Yao et al. (2002) | Q | 229 |
| | $6.7\times10^{-7}$ | | Yaws (1999) | ? | 21 |
| | $1.0\times10^{-6}$ | | Yaws and Yang (1992) | ? | 21 |
| 3-ethyl-2,2-dimethylhexane | $9.8\times10^{-7}$ | | Yaws (2003) | X | 237 |
| C$_{10}$H$_{22}$ | $1.6\times10^{-6}$ | | Gharagheizi et al. (2012) | Q | |
| [20291-91-2] | $1.1\times10^{-6}$ | | Gharagheizi et al. (2010) | Q | 246 |
| XYDYODCWVCBIOQ-UHFFFAOYSA-N | $1.8\times10^{-6}$ | | Hilal et al. (2008) | Q | |
| | $1.8\times10^{-6}$ | | Hilal et al. (2008) | Q | |
| | $1.3\times10^{-6}$ | | Modarresi et al. (2005) | Q | 247 |
| | $7.1\times10^{-7}$ | | Yao et al. (2002) | Q | 229 |
| | $9.8\times10^{-7}$ | | Yaws (1999) | ? | 21 |
| | $1.4\times10^{-6}$ | | Yaws and Yang (1992) | ? | 21 |
| 4-ethyl-2,2-dimethylhexane | $1.2\times10^{-6}$ | | Yaws (2003) | X | 237 |
| C$_{10}$H$_{22}$ | $1.2\times10^{-6}$ | | Gharagheizi et al. (2012) | Q | |
| [52896-99-8] | $1.1\times10^{-6}$ | | Gharagheizi et al. (2010) | Q | 246 |
| QHLDBFLIDFTHQI-UHFFFAOYSA-N | $1.1\times10^{-6}$ | | Hilal et al. (2008) | Q | |
| | $1.3\times10^{-6}$ | | Modarresi et al. (2005) | Q | 247 |
| | $9.9\times10^{-7}$ | | Yao et al. (2002) | Q | 229 |
| | $1.2\times10^{-6}$ | | Yaws (1999) | ? | 21 |
| | $1.6\times10^{-6}$ | | Yaws and Yang (1992) | ? | 21 |
| 3-ethyl-2,3-dimethylhexane | $8.5\times10^{-7}$ | | Yaws (2003) | X | 237 |
| C$_{10}$H$_{22}$ | $2.2\times10^{-6}$ | | Gharagheizi et al. (2012) | Q | |
| [52897-00-4] | $1.1\times10^{-6}$ | | Gharagheizi et al. (2010) | Q | 246 |
| PJIFKODHGMUPFH-UHFFFAOYSA-N | $2.0\times10^{-6}$ | | Hilal et al. (2008) | Q | |
| | $1.3\times10^{-6}$ | | Modarresi et al. (2005) | Q | 247 |
| | $6.6\times10^{-7}$ | | Yao et al. (2002) | Q | 229 |
| | $8.6\times10^{-7}$ | | Yaws (1999) | ? | 21 |
| | $1.3\times10^{-6}$ | | Yaws and Yang (1992) | ? | 21 |
| 4-ethyl-2,3-dimethylhexane | $9.4\times10^{-7}$ | | Yaws (2003) | X | 237 |
| C$_{10}$H$_{22}$ | $1.9\times10^{-6}$ | | Gharagheizi et al. (2012) | Q | |
| [52897-01-5] | $1.2\times10^{-6}$ | | Gharagheizi et al. (2010) | Q | 246 |
| RHMRCCBCYFAZIK-UHFFFAOYSA-N | $1.6\times10^{-6}$ | | Hilal et al. (2008) | Q | |
| | $1.4\times10^{-6}$ | | Modarresi et al. (2005) | Q | 247 |
| | $7.2\times10^{-7}$ | | Yao et al. (2002) | Q | 229 |
| | $9.5\times10^{-7}$ | | Yaws (1999) | ? | 21 |
| | $1.4\times10^{-6}$ | | Yaws and Yang (1992) | ? | 21 |



Table A2.1: Alkanes (... continued)

| Substance Formula (Trivial Name) [CAS Registry Number] InChIKey | $H_s^{cp}$ (at $T^{\ominus}$) $\left[\dfrac{\text{mol}}{\text{m}^3\,\text{Pa}}\right]$ | $\dfrac{\text{d}\ln H_s^{cp}}{\text{d}(1/T)}$ [K] | Reference | Type | Note |
|---|---|---|---|---|---|
| 3-ethyl-2,4-dimethylhexane $C_{10}H_{22}$ [7220-26-0] OSKIMJMPFNLVOU-UHFFFAOYSA-N | $9.5\times10^{-7}$ $1.9\times10^{-6}$ $1.2\times10^{-6}$ $1.6\times10^{-6}$ $1.3\times10^{-6}$ $7.1\times10^{-7}$ $9.5\times10^{-7}$ $1.4\times10^{-6}$ | | Yaws (2003) Gharagheizi et al. (2012) Gharagheizi et al. (2010) Hilal et al. (2008) Modarresi et al. (2005) Yao et al. (2002) Yaws (1999) Yaws and Yang (1992) | X Q Q Q Q Q ? ? | 237 246 247 229, 267 21 21 |
| 4-ethyl-2,4-dimethylhexane $C_{10}H_{22}$ [52897-03-7] SIKFMUYSQCEQOO-UHFFFAOYSA-N | $9.1\times10^{-7}$ $2.1\times10^{-6}$ $1.1\times10^{-6}$ $1.3\times10^{-6}$ $1.3\times10^{-6}$ $7.2\times10^{-7}$ $9.1\times10^{-7}$ $1.3\times10^{-6}$ | | Yaws (2003) Gharagheizi et al. (2012) Gharagheizi et al. (2010) Hilal et al. (2008) Modarresi et al. (2005) Yao et al. (2002) Yaws (1999) Yaws and Yang (1992) | X Q Q Q Q Q ? ? | 237 246 247 229 21 21 |
| 3-ethyl-2,5-dimethylhexane $C_{10}H_{22}$ [52897-04-8] UJEUVDLASLOZIV-UHFFFAOYSA-N | $1.1\times10^{-6}$ $1.5\times10^{-6}$ $1.2\times10^{-6}$ $1.1\times10^{-6}$ $1.3\times10^{-6}$ $1.1\times10^{-6}$ $1.1\times10^{-6}$ $1.5\times10^{-6}$ | | Yaws (2003) Gharagheizi et al. (2012) Gharagheizi et al. (2010) Hilal et al. (2008) Modarresi et al. (2005) Yao et al. (2002) Yaws (1999) Yaws and Yang (1992) | X Q Q Q Q Q ? ? | 237 246 247 229 21 21 |
| 4-ethyl-3,3-dimethylhexane $C_{10}H_{22}$ [52897-05-9] ZRTXVJYJVBTXHE-UHFFFAOYSA-N | $8.5\times10^{-7}$ $2.1\times10^{-6}$ $1.1\times10^{-6}$ $1.9\times10^{-6}$ $1.3\times10^{-6}$ $7.3\times10^{-7}$ $8.6\times10^{-7}$ $1.3\times10^{-6}$ | | Yaws (2003) Gharagheizi et al. (2012) Gharagheizi et al. (2010) Hilal et al. (2008) Modarresi et al. (2005) Yao et al. (2002) Yaws (1999) Yaws and Yang (1992) | X Q Q Q Q Q ? ? | 237 246 247 229 21 21 |
| 3-ethyl-3,4-dimethylhexane $C_{10}H_{22}$ [52897-06-0] ZGJCTUKRTSBTIQ-UHFFFAOYSA-N | $8.6\times10^{-7}$ $2.1\times10^{-6}$ $1.1\times10^{-6}$ $2.2\times10^{-6}$ $1.4\times10^{-6}$ $7.7\times10^{-7}$ $8.6\times10^{-7}$ $1.3\times10^{-6}$ | | Yaws (2003) Gharagheizi et al. (2012) Gharagheizi et al. (2010) Hilal et al. (2008) Modarresi et al. (2005) Yao et al. (2002) Yaws (1999) Yaws and Yang (1992) | X Q Q Q Q Q ? ? | 237 246 247 229 21 21 |



Table A2.1: Alkanes (. . . continued)

| Substance Formula (Trivial Name) [CAS Registry Number] InChIKey | $H_s^{cp}$ (at $T^\ominus$) $\left[ \dfrac{\text{mol}}{\text{m}^3\,\text{Pa}} \right]$ | $\dfrac{\text{d} \ln H_s^{cp}}{\text{d}(1/T)}$ [K] | Reference | Type | Note |
|---|---|---|---|---|---|
| 3,3-diethylhexane | $8.5 \times 10^{-7}$ | | Yaws (2003) | X | 237 |
| $C_{10}H_{22}$ | $2.3 \times 10^{-6}$ | | Gharagheizi et al. (2012) | Q | |
| [17302-02-2] | $1.1 \times 10^{-6}$ | | Gharagheizi et al. (2010) | Q | 246 |
| WWNGLKDLYKNGGT-UHFFFAOYSA-N | $3.4 \times 10^{-6}$ | | Hilal et al. (2008) | Q | |
| | $1.3 \times 10^{-6}$ | | Modarresi et al. (2005) | Q | 247 |
| | $7.7 \times 10^{-7}$ | | Yao et al. (2002) | Q | 229 |
| | $8.5 \times 10^{-7}$ | | Yaws (1999) | ? | 21 |
| | $1.3 \times 10^{-6}$ | | Yaws and Yang (1992) | ? | 21 |
| 3,4-diethylhexane | $9.3 \times 10^{-7}$ | | Yaws (2003) | X | 237 |
| $C_{10}H_{22}$ | $2.1 \times 10^{-6}$ | | Gharagheizi et al. (2012) | Q | |
| [19398-77-7] | $1.2 \times 10^{-6}$ | | Gharagheizi et al. (2010) | Q | 246 |
| VBZCRMTUDYIWIH-UHFFFAOYSA-N | $1.8 \times 10^{-6}$ | | Hilal et al. (2008) | Q | |
| | $1.3 \times 10^{-6}$ | | Modarresi et al. (2005) | Q | 247 |
| | $7.9 \times 10^{-7}$ | | Yao et al. (2002) | Q | 229 |
| | $9.3 \times 10^{-7}$ | | Yaws (1999) | ? | 21 |
| | $1.4 \times 10^{-6}$ | | Yaws and Yang (1992) | ? | 21 |
| 2-methyl-3-(1-methylethyl)-hexane | $7.5 \times 10^{-7}$ | | Yaws (2003) | X | 237 |
| $C_{10}H_{22}$ | $2.2 \times 10^{-6}$ | | Gharagheizi et al. (2012) | Q | |
| (3-isopropyl-2-methylhexane) | $1.2 \times 10^{-6}$ | | Gharagheizi et al. (2010) | Q | 246 |
| [62016-13-1] | $1.5 \times 10^{-6}$ | | Hilal et al. (2008) | Q | |
| YBOXGRMAQIYMGV-UHFFFAOYSA-N | $1.4 \times 10^{-6}$ | | Modarresi et al. (2005) | Q | 247 |
| | $8.5 \times 10^{-7}$ | | Yao et al. (2002) | Q | 229, 267 |
| | $7.5 \times 10^{-7}$ | | Yaws (1999) | ? | 21 |
| | $1.1 \times 10^{-6}$ | | Yaws and Yang (1992) | ? | 21 |
| 2,2,3,3,4-pentamethylpentane | $6.8 \times 10^{-7}$ | | Yaws (2003) | X | 237 |
| $C_{10}H_{22}$ | $2.6 \times 10^{-6}$ | | Gharagheizi et al. (2012) | Q | |
| [16747-44-7] | $1.0 \times 10^{-6}$ | | Gharagheizi et al. (2010) | Q | 246 |
| WKQBIIUOSATALN-UHFFFAOYSA-N | $1.3 \times 10^{-6}$ | | Hilal et al. (2008) | Q | |
| | $1.4 \times 10^{-6}$ | | Modarresi et al. (2005) | Q | 247 |
| | $8.8 \times 10^{-7}$ | | Yao et al. (2002) | Q | 229, 267 |
| | $6.8 \times 10^{-7}$ | | Yaws (1999) | ? | 21 |
| | $1.0 \times 10^{-6}$ | | Yaws and Yang (1992) | ? | 21 |
| 2,2,3,4,4-pentamethylpentane | $7.2 \times 10^{-7}$ | | Yaws (2003) | X | 237 |
| $C_{10}H_{22}$ | $1.9 \times 10^{-6}$ | | Gharagheizi et al. (2012) | Q | |
| [16747-45-8] | $1.0 \times 10^{-6}$ | | Gharagheizi et al. (2010) | Q | 246 |
| OWFKEHICSVOVAC-UHFFFAOYSA-N | $8.6 \times 10^{-7}$ | | Hilal et al. (2008) | Q | |
| | $1.4 \times 10^{-6}$ | | Modarresi et al. (2005) | Q | 247 |
| | $1.0 \times 10^{-6}$ | | Yao et al. (2002) | Q | 229 |
| | $7.2 \times 10^{-7}$ | | Yaws (1999) | ? | 21 |
| | $1.0 \times 10^{-6}$ | | Yaws and Yang (1992) | ? | 21 |



Table A2.1: Alkanes (. . . continued)

| Substance Formula (Trivial Name) [CAS Registry Number] InChIKey | $H_s^{cp}$ (at $T^\ominus$) $\left[\dfrac{\text{mol}}{\text{m}^3\,\text{Pa}}\right]$ | $\dfrac{\text{d}\ln H_s^{cp}}{\text{d}(1/T)}$ [K] | Reference | Type | Note |
|---|---|---|---|---|---|
| 3-ethyl-2,2,3-trimethylpentane $C_{10}H_{22}$ [52897-17-3] AJDIFHIHSYVDGP-UHFFFAOYSA-N | $6.6\times10^{-7}$ $2.8\times10^{-6}$ $9.9\times10^{-7}$ $2.1\times10^{-6}$ $1.4\times10^{-6}$ $9.2\times10^{-7}$ $6.6\times10^{-7}$ $1.0\times10^{-6}$ | | Yaws (2003) Gharagheizi et al. (2012) Gharagheizi et al. (2010) Hilal et al. (2008) Modarresi et al. (2005) Yao et al. (2002) Yaws (1999) Yaws and Yang (1992) | X Q Q Q Q Q ? ? | 237 246 247 229 21 21 |
| 3-ethyl-2,2,4-trimethylpentane $C_{10}H_{22}$ [52897-18-4] VLIZIVHXZXQRDE-UHFFFAOYSA-N | $9.2\times10^{-7}$ $1.7\times10^{-6}$ $1.1\times10^{-6}$ $1.2\times10^{-6}$ $1.4\times10^{-6}$ $1.1\times10^{-6}$ $9.2\times10^{-7}$ $1.3\times10^{-6}$ | | Yaws (2003) Gharagheizi et al. (2012) Gharagheizi et al. (2010) Hilal et al. (2008) Modarresi et al. (2005) Yao et al. (2002) Yaws (1999) Yaws and Yang (1992) | X Q Q Q Q Q ? ? | 237 246 247 229 21 21 |
| 3-ethyl-2,3,4-trimethylpentane $C_{10}H_{22}$ [52897-19-5] OHZNMGSGEFVFTI-UHFFFAOYSA-N | $7.0\times10^{-7}$ $2.8\times10^{-6}$ $1.1\times10^{-6}$ $1.5\times10^{-6}$ $1.3\times10^{-6}$ $1.0\times10^{-6}$ $7.0\times10^{-7}$ $1.1\times10^{-6}$ | | Yaws (2003) Gharagheizi et al. (2012) Gharagheizi et al. (2010) Hilal et al. (2008) Modarresi et al. (2005) Yao et al. (2002) Yaws (1999) Yaws and Yang (1992) | X Q Q Q Q Q ? ? | 237 246 247 229 21 21 |
| 3,3-diethyl-2-methylpentane $C_{10}H_{22}$ [52897-16-2] DSSAZLXYIQIXGW-UHFFFAOYSA-N | $7.5\times10^{-7}$ $2.8\times10^{-6}$ $1.1\times10^{-6}$ $2.3\times10^{-6}$ $1.4\times10^{-6}$ $1.2\times10^{-6}$ $7.5\times10^{-7}$ $1.1\times10^{-6}$ | | Yaws (2003) Gharagheizi et al. (2012) Gharagheizi et al. (2010) Hilal et al. (2008) Modarresi et al. (2005) Yao et al. (2002) Yaws (1999) Yaws and Yang (1992) | X Q Q Q Q Q ? ? | 237 246 247 229 21 21 |
| 2,4-dimethyl-3-(1-methylethyl)-pentane $C_{10}H_{22}$ (2,4-dimethyl-3-isopropylpentane) [13475-79-1] YVYHOOYMDHZALB-UHFFFAOYSA-N | $9.1\times10^{-7}$ $1.7\times10^{-6}$ $1.2\times10^{-6}$ $1.0\times10^{-6}$ $1.3\times10^{-6}$ $1.2\times10^{-6}$ $9.2\times10^{-7}$ $1.3\times10^{-6}$ | | Yaws (2003) Gharagheizi et al. (2012) Gharagheizi et al. (2010) Hilal et al. (2008) Modarresi et al. (2005) Yao et al. (2002) Yaws (1999) Yaws and Yang (1992) | X Q Q Q Q Q ? ? | 237 246 247 229 21 21 |



Table A2.1: Alkanes (...continued)

| Substance Formula (Trivial Name) [CAS Registry Number] InChIKey | $H_s^{cp}$ (at $T^{\ominus}$) $\left[\dfrac{\mathrm{mol}}{\mathrm{m^3\,Pa}}\right]$ | $\dfrac{\mathrm{d}\ln H_s^{cp}}{\mathrm{d}(1/T)}$ [K] | Reference | Type | Note |
|---|---|---|---|---|---|
| undecane | $4.5\times10^{-7}$ | | Plyasunov and Shock (2000) | L | |
| $C_{11}H_{24}$ | $5.4\times10^{-7}$ | | Mackay and Shiu (1981) | L | |
| [1120-21-4] | $5.1\times10^{-6}$ | | Duchowicz et al. (2020) | V | 186 |
| RSJKGSCJYJTIGS-UHFFFAOYSA-N | $5.2\times10^{-6}$ | | HSDB (2015) | V | |
| | $4.9\times10^{-7}$ | | Mackay et al. (2006a) | V | |
| | $5.4\times10^{-6}$ | | Eastcott et al. (1988) | V | |
| | $4.9\times10^{-7}$ | | Abraham (1984) | V | |
| | $5.1\times10^{-6}$ | | Yaws (2003) | X | 237 |
| | $4.4\times10^{-4}$ | | Duchowicz et al. (2020) | Q | |
| | $2.1\times10^{-5}$ | | Wang et al. (2017) | Q | 80, 238 |
| | $1.2\times10^{-6}$ | | Wang et al. (2017) | Q | 80, 239 |
| | $1.1\times10^{-5}$ | | Wang et al. (2017) | Q | 80, 240 |
| | $4.4\times10^{-6}$ | | Gharagheizi et al. (2012) | Q | |
| | $1.5\times10^{-6}$ | | Gharagheizi et al. (2010) | Q | 246 |
| | $1.5\times10^{-6}$ | | Hilal et al. (2008) | Q | |
| | $3.6\times10^{-6}$ | | Modarresi et al. (2007) | Q | 67 |
| | $5.4\times10^{-7}$ | | Yaffe et al. (2003) | Q | 248, 249 |
| | $3.4\times10^{-6}$ | | Yao et al. (2002) | Q | 229, 267 |
| | $5.1\times10^{-6}$ | | Yaws (1999) | ? | 21 |
| | $5.4\times10^{-6}$ | | Yaws and Yang (1992) | ? | 21 |
| 2-methyldecane | $1.1\times10^{-6}$ | | Yaws (2003) | X | 237 |
| $C_{11}H_{24}$ | $3.6\times10^{-6}$ | | Gharagheizi et al. (2012) | Q | |
| [6975-98-0] | $1.2\times10^{-6}$ | | Gharagheizi et al. (2010) | Q | 246 |
| CNPVJWYWYZMPDS-UHFFFAOYSA-N | | | | | |
| 3-methyldecane | $1.1\times10^{-6}$ | | Yaws (2003) | X | 237 |
| $C_{11}H_{24}$ | $2.4\times10^{-6}$ | | Gharagheizi et al. (2012) | Q | |
| [13151-34-3] | $1.2\times10^{-6}$ | | Gharagheizi et al. (2010) | Q | 246 |
| JJRUZTXRDDMYGM-UHFFFAOYSA-N | | | | | |
| 4-methyldecane | $1.1\times10^{-6}$ | | Yaws (2003) | X | 237 |
| $C_{11}H_{24}$ | $2.3\times10^{-6}$ | | Gharagheizi et al. (2012) | Q | |
| [2847-72-5] | $1.2\times10^{-6}$ | | Gharagheizi et al. (2010) | Q | 246 |
| DVWZNKLWPILULD-UHFFFAOYSA-N | | | | | |
| 5-methyldecane | $1.1\times10^{-6}$ | | Yaws (2003) | X | 237 |
| $C_{11}H_{24}$ | $2.2\times10^{-6}$ | | Gharagheizi et al. (2012) | Q | |
| [13151-35-4] | $1.2\times10^{-6}$ | | Gharagheizi et al. (2010) | Q | 246 |
| QUYFPNWYGLFQQU-UHFFFAOYSA-N | | | | | |
| 2,2-dimethylnonane | $9.7\times10^{-7}$ | | Yaws (2003) | X | 237 |
| $C_{11}H_{24}$ | $1.2\times10^{-6}$ | | Gharagheizi et al. (2012) | Q | |
| [17302-14-6] | $1.0\times10^{-6}$ | | Gharagheizi et al. (2010) | Q | 246 |
| WDSBVMLUILIJOW-UHFFFAOYSA-N | | | | | |
| 2,3-dimethylnonane | $1.0\times10^{-6}$ | | Yaws (2003) | X | 237 |
| $C_{11}H_{24}$ | $1.4\times10^{-6}$ | | Gharagheizi et al. (2012) | Q | |
| [2884-06-2] | $1.0\times10^{-6}$ | | Gharagheizi et al. (2010) | Q | 246 |
| IGJRNTLDWLTHCQ-UHFFFAOYSA-N | | | | | |





Table A2.1: Alkanes (. . . continued)

| Substance<br>Formula<br>(Trivial Name)<br>[CAS Registry Number]<br>InChIKey | $H_s^{cp}$<br>(at $T^{\ominus}$)<br>$\left[\dfrac{\mathrm{mol}}{\mathrm{m}^3\,\mathrm{Pa}}\right]$ | $\dfrac{\mathrm{d}\ln H_s^{cp}}{\mathrm{d}(1/T)}$<br><br>[K] | Reference | Type | Note |
|---|---|---|---|---|---|
| 2,4-dimethylnonane<br>C$_{11}$H$_{24}$<br>[17302-24-8]<br>JZUUOAUSQCXSTN-UHFFFAOYSA-N | $1.1\times10^{-6}$<br>$1.1\times10^{-6}$<br>$1.0\times10^{-6}$ | | Yaws (2003)<br>Gharagheizi et al. (2012)<br>Gharagheizi et al. (2010) | X<br>Q<br>Q | 237<br><br>246 |
| 2,5-dimethylnonane<br>C$_{11}$H$_{24}$<br>[17302-27-1]<br>NQUMJENPNGXAIH-UHFFFAOYSA-N | $1.1\times10^{-6}$<br>$1.2\times10^{-6}$<br>$1.0\times10^{-6}$ | | Yaws (2003)<br>Gharagheizi et al. (2012)<br>Gharagheizi et al. (2010) | X<br>Q<br>Q | 237<br><br>246 |
| 2,6-dimethylnonane<br>C$_{11}$H$_{24}$<br>[17302-28-2]<br>MGNGOEWESNDQAN-UHFFFAOYSA-N | $1.1\times10^{-6}$<br>$1.2\times10^{-6}$<br>$1.0\times10^{-6}$ | | Yaws (2003)<br>Gharagheizi et al. (2012)<br>Gharagheizi et al. (2010) | X<br>Q<br>Q | 237<br><br>246 |
| 2,7-dimethylnonane<br>C$_{11}$H$_{24}$<br>[17302-29-3]<br>QYQSPINNJUXEDY-UHFFFAOYSA-N | $1.1\times10^{-6}$<br>$1.2\times10^{-6}$<br>$1.0\times10^{-6}$ | | Yaws (2003)<br>Gharagheizi et al. (2012)<br>Gharagheizi et al. (2010) | X<br>Q<br>Q | 237<br><br>246 |
| 2,8-dimethylnonane<br>C$_{11}$H$_{24}$<br>[17302-30-6]<br>FZFRYQHIQUEAAV-UHFFFAOYSA-N | $1.0\times10^{-6}$<br>$1.3\times10^{-6}$<br>$1.0\times10^{-6}$ | | Yaws (2003)<br>Gharagheizi et al. (2012)<br>Gharagheizi et al. (2010) | X<br>Q<br>Q | 237<br><br>246 |
| 3,3-dimethylnonane<br>C$_{11}$H$_{24}$<br>[17302-15-7]<br>HTRYNYZYFGHKDV-UHFFFAOYSA-N | $9.5\times10^{-7}$<br>$1.3\times10^{-6}$<br>$1.0\times10^{-6}$ | | Yaws (2003)<br>Gharagheizi et al. (2012)<br>Gharagheizi et al. (2010) | X<br>Q<br>Q | 237<br><br>246 |
| 3,4-dimethylnonane<br>C$_{11}$H$_{24}$<br>[17302-22-6]<br>PAXLEVPLFYBSQJ-UHFFFAOYSA-N | $1.0\times10^{-6}$<br>$1.4\times10^{-6}$<br>$1.0\times10^{-6}$ | | Yaws (2003)<br>Gharagheizi et al. (2012)<br>Gharagheizi et al. (2010) | X<br>Q<br>Q | 237<br><br>246 |
| 3,5-dimethylnonane<br>C$_{11}$H$_{24}$<br>[17302-25-9]<br>BAFVBVRBYKSWCE-UHFFFAOYSA-N | $1.1\times10^{-6}$<br>$1.2\times10^{-6}$<br>$1.0\times10^{-6}$ | | Yaws (2003)<br>Gharagheizi et al. (2012)<br>Gharagheizi et al. (2010) | X<br>Q<br>Q | 237<br><br>246 |
| 3,6-dimethylnonane<br>C$_{11}$H$_{24}$<br>[17302-31-7]<br>YHLBUWVGPXILSW-UHFFFAOYSA-N | $1.0\times10^{-6}$<br>$1.3\times10^{-6}$<br>$1.0\times10^{-6}$ | | Yaws (2003)<br>Gharagheizi et al. (2012)<br>Gharagheizi et al. (2010) | X<br>Q<br>Q | 237<br><br>246 |
| 3,7-dimethylnonane<br>C$_{11}$H$_{24}$<br>[17302-32-8]<br>YGPVLXJHRFZYJJ-UHFFFAOYSA-N | $1.0\times10^{-6}$<br>$1.4\times10^{-6}$<br>$1.0\times10^{-6}$ | | Yaws (2003)<br>Gharagheizi et al. (2012)<br>Gharagheizi et al. (2010) | X<br>Q<br>Q | 237<br><br>246 |



Table A2.1: Alkanes (... continued)

| Substance<br>Formula<br>(Trivial Name)<br>[CAS Registry Number]<br>InChIKey | $H_s^{cp}$<br>(at $T^\ominus$)<br>$\left[\dfrac{\mathrm{mol}}{\mathrm{m^3\,Pa}}\right]$ | $\dfrac{\mathrm{d\ln} H_s^{cp}}{\mathrm{d}(1/T)}$<br><br>[K] | Reference | Type | Note |
|---|---|---|---|---|---|
| 4,4-dimethylnonane<br>$C_{11}H_{24}$<br>[17302-18-0]<br>HARRKRKVTQABSF-UHFFFAOYSA-N | $9.9\times10^{-7}$<br>$1.1\times10^{-6}$<br>$1.0\times10^{-6}$ | | Yaws (2003)<br>Gharagheizi et al. (2012)<br>Gharagheizi et al. (2010) | X<br>Q<br>Q | 237<br><br>246 |
| 4,5-dimethylnonane<br>$C_{11}H_{24}$<br>[17302-23-7]<br>JDNGDDOTBYZAGS-UHFFFAOYSA-N | $1.0\times10^{-6}$<br>$1.3\times10^{-6}$<br>$1.0\times10^{-6}$ | | Yaws (2003)<br>Gharagheizi et al. (2012)<br>Gharagheizi et al. (2010) | X<br>Q<br>Q | 237<br><br>246 |
| 4,6-dimethylnonane<br>$C_{11}H_{24}$<br>[17302-26-0]<br>JZKWOUUZMBTBDO-UHFFFAOYSA-N | $1.1\times10^{-6}$<br>$1.2\times10^{-6}$<br>$1.0\times10^{-6}$ | | Yaws (2003)<br>Gharagheizi et al. (2012)<br>Gharagheizi et al. (2010) | X<br>Q<br>Q | 237<br><br>246 |
| 5,5-dimethylnonane<br>$C_{11}H_{24}$<br>[6414-96-6]<br>BEPKYSJVUZWKMP-UHFFFAOYSA-N | $1.0\times10^{-6}$<br>$1.1\times10^{-6}$<br>$1.0\times10^{-6}$ | | Yaws (2003)<br>Gharagheizi et al. (2012)<br>Gharagheizi et al. (2010) | X<br>Q<br>Q | 237<br><br>246 |
| 3-ethylnonane<br>$C_{11}H_{24}$<br>[17302-11-3]<br>FKJSIWPQZCKMIL-UHFFFAOYSA-N | $1.1\times10^{-6}$<br>$1.5\times10^{-6}$<br>$1.2\times10^{-6}$ | | Yaws (2003)<br>Gharagheizi et al. (2012)<br>Gharagheizi et al. (2010) | X<br>Q<br>Q | 237<br><br>246 |
| 4-ethylnonane<br>$C_{11}H_{24}$<br>[5911-05-7]<br>UGCQDCMVAKKTQG-UHFFFAOYSA-N | $1.1\times10^{-6}$<br>$1.3\times10^{-6}$<br>$1.2\times10^{-6}$ | | Yaws (2003)<br>Gharagheizi et al. (2012)<br>Gharagheizi et al. (2010) | X<br>Q<br>Q | 237<br><br>246 |
| 5-ethylnonane<br>$C_{11}H_{24}$<br>[17302-12-4]<br>QPOVZYGNFWRMJE-UHFFFAOYSA-N | $1.1\times10^{-6}$<br>$1.2\times10^{-6}$<br>$1.2\times10^{-6}$ | | Yaws (2003)<br>Gharagheizi et al. (2012)<br>Gharagheizi et al. (2010) | X<br>Q<br>Q | 237<br><br>246 |
| 2,2,3-trimethyloctane<br>$C_{11}H_{24}$<br>[62016-26-6]<br>XZPXMWMVWDSEMY-UHFFFAOYSA-N | $9.0\times10^{-7}$<br>$1.2\times10^{-6}$<br>$9.0\times10^{-7}$ | | Yaws (2003)<br>Gharagheizi et al. (2012)<br>Gharagheizi et al. (2010) | X<br>Q<br>Q | 237<br><br>246 |
| 2,2,4-trimethyloctane<br>$C_{11}H_{24}$<br>[18932-14-4]<br>IKMGZPRUMVFYBK-UHFFFAOYSA-N | $9.9\times10^{-7}$<br>$9.6\times10^{-7}$<br>$9.0\times10^{-7}$ | | Yaws (2003)<br>Gharagheizi et al. (2012)<br>Gharagheizi et al. (2010) | X<br>Q<br>Q | 237<br><br>246 |
| 2,2,5-trimethyloctane<br>$C_{11}H_{24}$<br>[62016-27-7]<br>CSVNISDVBYGXME-UHFFFAOYSA-N | $9.9\times10^{-7}$<br>$9.5\times10^{-7}$<br>$9.0\times10^{-7}$ | | Yaws (2003)<br>Gharagheizi et al. (2012)<br>Gharagheizi et al. (2010) | X<br>Q<br>Q | 237<br><br>246 |



Table A2.1: Alkanes (. . . continued)

| Substance Formula (Trivial Name) [CAS Registry Number] InChIKey | $H_s^{cp}$ (at $T^\ominus$) $\left[\dfrac{\text{mol}}{\text{m}^3\,\text{Pa}}\right]$ | $\dfrac{\mathrm{d}\ln H_s^{cp}}{\mathrm{d}(1/T)}$ [K] | Reference | Type | Note |
|---|---|---|---|---|---|
| 2,2,6-trimethyloctane $C_{11}H_{24}$ [62016-28-8] NBIHFQKVSFKHGH-UHFFFAOYSA-N | $9.6\times10^{-7}$ $1.0\times10^{-6}$ $9.0\times10^{-7}$ | | Yaws (2003) Gharagheizi et al. (2012) Gharagheizi et al. (2010) | X Q Q | 237  246 |
| 2,2,7-trimethyloctane $C_{11}H_{24}$ [62016-29-9] QGQAHVJOBYNMFN-UHFFFAOYSA-N | $1.0\times10^{-6}$ $9.2\times10^{-7}$ $9.0\times10^{-7}$ | | Yaws (2003) Gharagheizi et al. (2012) Gharagheizi et al. (2010) | X Q Q | 237  246 |
| 2,3,3-trimethyloctane $C_{11}H_{24}$ [62016-30-2] KFYWDCDWVPXXCA-UHFFFAOYSA-N | $8.7\times10^{-7}$ $1.3\times10^{-6}$ $9.0\times10^{-7}$ | | Yaws (2003) Gharagheizi et al. (2012) Gharagheizi et al. (2010) | X Q Q | 237  246 |
| 2,3,4-trimethyloctane $C_{11}H_{24}$ [62016-31-3] SJQAEXCGOLTHPG-UHFFFAOYSA-N | $9.7\times10^{-7}$ $1.2\times10^{-6}$ $9.7\times10^{-7}$ | | Yaws (2003) Gharagheizi et al. (2012) Gharagheizi et al. (2010) | X Q Q | 237  246 |
| 2,3,5-trimethyloctane $C_{11}H_{24}$ [62016-32-4] CEOHXVQAHFSSEG-UHFFFAOYSA-N | $1.0\times10^{-6}$ $1.1\times10^{-6}$ $9.7\times10^{-7}$ | | Yaws (2003) Gharagheizi et al. (2012) Gharagheizi et al. (2010) | X Q Q | 237  246 |
| 2,3,6-trimethyloctane $C_{11}H_{24}$ [62016-33-5] MNFBUNLEXRMWOY-UHFFFAOYSA-N | $9.7\times10^{-7}$ $1.2\times10^{-6}$ $9.7\times10^{-7}$ | | Yaws (2003) Gharagheizi et al. (2012) Gharagheizi et al. (2010) | X Q Q | 237  246 |
| 2,3,7-trimethyloctane $C_{11}H_{24}$ [62016-34-6] XJKKSYAVEVAGFX-UHFFFAOYSA-N | $9.8\times10^{-7}$ $1.2\times10^{-6}$ $9.7\times10^{-7}$ | | Yaws (2003) Gharagheizi et al. (2012) Gharagheizi et al. (2010) | X Q Q | 237  246 |
| 2,4,4-trimethyloctane $C_{11}H_{24}$ [62016-35-7] CJXXTMYWNCYKHU-UHFFFAOYSA-N | $9.9\times10^{-7}$ $9.5\times10^{-7}$ $9.0\times10^{-7}$ | | Yaws (2003) Gharagheizi et al. (2012) Gharagheizi et al. (2010) | X Q Q | 237  246 |
| 2,4,5-trimethyloctane $C_{11}H_{24}$ [62016-36-8] UJYGOBMOHSFJQP-UHFFFAOYSA-N | $1.0\times10^{-6}$ $1.1\times10^{-6}$ $9.7\times10^{-7}$ | | Yaws (2003) Gharagheizi et al. (2012) Gharagheizi et al. (2010) | X Q Q | 237  246 |
| 2,4,6-trimethyloctane $C_{11}H_{24}$ [62016-37-9] XHNIFDXYGLPJLP-UHFFFAOYSA-N | $1.0\times10^{-6}$ $1.0\times10^{-6}$ $9.7\times10^{-7}$ | | Yaws (2003) Gharagheizi et al. (2012) Gharagheizi et al. (2010) | X Q Q | 237  246 |





Table A2.1: Alkanes (...continued)

| Substance Formula (Trivial Name) [CAS Registry Number] InChIKey | $H_s^{cp}$ (at $T^{\ominus}$) $\left[\dfrac{\text{mol}}{\text{m}^3\,\text{Pa}}\right]$ | $\dfrac{\text{d}\ln H_s^{cp}}{\text{d}(1/T)}$ [K] | Reference | Type | Note |
|---|---|---|---|---|---|
| 2,4,7-trimethyloctane | $1.1\times10^{-6}$ | | Yaws (2003) | X | 237 |
| $C_{11}H_{24}$ | $9.4\times10^{-7}$ | | Gharagheizi et al. (2012) | Q | |
| [62016-38-0] | $9.7\times10^{-7}$ | | Gharagheizi et al. (2010) | Q | 246 |
| DUHKHXHIPOGMOW-UHFFFAOYSA-N | | | | | |
| 2,5,5-trimethyloctane | $9.8\times10^{-7}$ | | Yaws (2003) | X | 237 |
| $C_{11}H_{24}$ | $9.7\times10^{-7}$ | | Gharagheizi et al. (2012) | Q | |
| [62016-39-1] | $9.0\times10^{-7}$ | | Gharagheizi et al. (2010) | Q | 246 |
| DZKJZWAIBSEZKB-UHFFFAOYSA-N | | | | | |
| 2,5,6-trimethyloctane | $1.0\times10^{-6}$ | | Yaws (2003) | X | 237 |
| $C_{11}H_{24}$ | $1.2\times10^{-6}$ | | Gharagheizi et al. (2012) | Q | |
| [62016-14-2] | $9.7\times10^{-7}$ | | Gharagheizi et al. (2010) | Q | 246 |
| VHBZECSWMWWTMQ-UHFFFAOYSA-N | | | | | |
| 2,6,6-trimethyloctane | $9.3\times10^{-7}$ | | Yaws (2003) | X | 237 |
| $C_{11}H_{24}$ | $1.1\times10^{-6}$ | | Gharagheizi et al. (2012) | Q | |
| [54166-32-4] | $9.0\times10^{-7}$ | | Gharagheizi et al. (2010) | Q | 246 |
| RUPXAIGHLDMSOL-UHFFFAOYSA-N | | | | | |
| 3,3,4-trimethyloctane | $8.7\times10^{-7}$ | | Yaws (2003) | X | 237 |
| $C_{11}H_{24}$ | $1.3\times10^{-6}$ | | Gharagheizi et al. (2012) | Q | |
| [62016-40-4] | $9.0\times10^{-7}$ | | Gharagheizi et al. (2010) | Q | 246 |
| IJVIFYHTPQSPSW-UHFFFAOYSA-N | | | | | |
| 3,3,5-trimethyloctane | $9.6\times10^{-7}$ | | Yaws (2003) | X | 237 |
| $C_{11}H_{24}$ | $1.0\times10^{-6}$ | | Gharagheizi et al. (2012) | Q | |
| [62016-41-5] | $9.0\times10^{-7}$ | | Gharagheizi et al. (2010) | Q | 246 |
| BEFCOJKHEMBPSK-UHFFFAOYSA-N | | | | | |
| 3,3,6-trimethyloctane | $9.2\times10^{-7}$ | | Yaws (2003) | X | 237 |
| $C_{11}H_{24}$ | $1.1\times10^{-6}$ | | Gharagheizi et al. (2012) | Q | |
| [62016-42-6] | $9.0\times10^{-7}$ | | Gharagheizi et al. (2010) | Q | 246 |
| CFESHXNQRRYSED-UHFFFAOYSA-N | | | | | |
| 3,4,4-trimethyloctane | $8.8\times10^{-7}$ | | Yaws (2003) | X | 237 |
| $C_{11}H_{24}$ | $1.3\times10^{-6}$ | | Gharagheizi et al. (2012) | Q | |
| [62016-43-7] | $9.0\times10^{-7}$ | | Gharagheizi et al. (2010) | Q | 246 |
| WWCBWKZBTINVDE-UHFFFAOYSA-N | | | | | |
| 3,4,5-trimethyloctane | $9.6\times10^{-7}$ | | Yaws (2003) | X | 237 |
| $C_{11}H_{24}$ | $1.3\times10^{-6}$ | | Gharagheizi et al. (2012) | Q | |
| [62016-44-8] | $9.7\times10^{-7}$ | | Gharagheizi et al. (2010) | Q | 246 |
| BPFOUTQNEVQSFR-UHFFFAOYSA-N | | | | | |
| 3,4,6-trimethyloctane | $9.8\times10^{-7}$ | | Yaws (2003) | X | 237 |
| $C_{11}H_{24}$ | $1.2\times10^{-6}$ | | Gharagheizi et al. (2012) | Q | |
| [62016-45-9] | $9.7\times10^{-7}$ | | Gharagheizi et al. (2010) | Q | 246 |
| RMGAENUTYIFEJO-UHFFFAOYSA-N | | | | | |



Table A2.1: Alkanes (. . . continued)

| Substance Formula (Trivial Name) [CAS Registry Number] InChIKey | $H_s^{cp}$ (at $T^\ominus$) $\left[\dfrac{\mathrm{mol}}{\mathrm{m}^3\,\mathrm{Pa}}\right]$ | $\dfrac{\mathrm{d}\ln H_s^{cp}}{\mathrm{d}(1/T)}$ [K] | Reference | Type | Note |
|---|---|---|---|---|---|
| 3,5,5-trimethyloctane $C_{11}H_{24}$ [61868-94-8] FVYNWISHVUNIOZ-UHFFFAOYSA-N | $1.0\times10^{-6}$ $9.2\times10^{-7}$ $9.0\times10^{-7}$ | | Yaws (2003) Gharagheizi et al. (2012) Gharagheizi et al. (2010) | X Q Q | 237 246 |
| 4,4,5-trimethyloctane $C_{11}H_{24}$ [61868-95-9] QGCXVKCDWOOWTQ-UHFFFAOYSA-N | $8.9\times10^{-7}$ $1.2\times10^{-6}$ $9.0\times10^{-7}$ | | Yaws (2003) Gharagheizi et al. (2012) Gharagheizi et al. (2010) | X Q Q | 237 246 |
| 2-methyl-3-ethyloctane $C_{11}H_{24}$ [62016-16-4] ITDXDXCBDQBZEB-UHFFFAOYSA-N | $1.0\times10^{-6}$ $1.3\times10^{-6}$ $1.0\times10^{-6}$ | | Yaws (2003) Gharagheizi et al. (2012) Gharagheizi et al. (2010) | X Q Q | 237 246 |
| 2-methyl-4-ethyloctane $C_{11}H_{24}$ [62016-17-5] ZYEQSLLHBOITAM-UHFFFAOYSA-N | $1.1\times10^{-6}$ $1.1\times10^{-6}$ $1.0\times10^{-6}$ | | Yaws (2003) Gharagheizi et al. (2012) Gharagheizi et al. (2010) | X Q Q | 237 246 |
| 2-methyl-5-ethyloctane $C_{11}H_{24}$ [62016-18-6] CQCKNPUKBOITAX-UHFFFAOYSA-N | $1.1\times10^{-6}$ $1.1\times10^{-6}$ $1.0\times10^{-6}$ | | Yaws (2003) Gharagheizi et al. (2012) Gharagheizi et al. (2010) | X Q Q | 237 246 |
| 2-methyl-6-ethyloctane $C_{11}H_{24}$ [62016-19-7] AZXGABNJUBNOHW-UHFFFAOYSA-N | $1.0\times10^{-6}$ $1.3\times10^{-6}$ $1.0\times10^{-6}$ | | Yaws (2003) Gharagheizi et al. (2012) Gharagheizi et al. (2010) | X Q Q | 237 246 |
| 3-methyl-3-ethyloctane $C_{11}H_{24}$ [17302-16-8] DQNINFLTCGTQGU-UHFFFAOYSA-N | $9.1\times10^{-7}$ $1.4\times10^{-6}$ $1.0\times10^{-6}$ | | Yaws (2003) Gharagheizi et al. (2012) Gharagheizi et al. (2010) | X Q Q | 237 246 |
| 3-methyl-4-ethyloctane $C_{11}H_{24}$ [62016-20-0] BTRAURWAFYPYMW-UHFFFAOYSA-N | $1.0\times10^{-6}$ $1.3\times10^{-6}$ $1.0\times10^{-6}$ | | Yaws (2003) Gharagheizi et al. (2012) Gharagheizi et al. (2010) | X Q Q | 237 246 |
| 3-methyl-5-ethyloctane $C_{11}H_{24}$ [62016-21-1] LZFCEXJTLLKGPW-UHFFFAOYSA-N | $1.1\times10^{-6}$ $1.1\times10^{-6}$ $1.0\times10^{-6}$ | | Yaws (2003) Gharagheizi et al. (2012) Gharagheizi et al. (2010) | X Q Q | 237 246 |
| 3-methyl-6-ethyloctane $C_{11}H_{24}$ [62016-22-2] FTQLPWORENXYAZ-UHFFFAOYSA-N | $1.0\times10^{-6}$ $1.3\times10^{-6}$ $1.0\times10^{-6}$ | | Yaws (2003) Gharagheizi et al. (2012) Gharagheizi et al. (2010) | X Q Q | 237 246 |





Table A2.1: Alkanes (...continued)

| Substance Formula (Trivial Name) [CAS Registry Number] InChIKey | $H_s^{cp}$ (at $T^{\ominus}$) $\left[\dfrac{\text{mol}}{\text{m}^3\,\text{Pa}}\right]$ | $\dfrac{\text{d}\ln H_s^{cp}}{\text{d}(1/T)}$ [K] | Reference | Type | Note |
|---|---|---|---|---|---|
| 4-methyl-3-ethyloctane $C_{11}H_{24}$ [62016-23-3] LXRSBMFQRZTMNK-UHFFFAOYSA-N | $1.0\times10^{-6}$ $1.3\times10^{-6}$ $1.0\times10^{-6}$ | | Yaws (2003) Gharagheizi et al. (2012) Gharagheizi et al. (2010) | X Q Q | 237 246 |
| 4-methyl-4-ethyloctane $C_{11}H_{24}$ [17302-19-1] IXZULXYHNRHENR-UHFFFAOYSA-N | $9.7\times10^{-7}$ $1.2\times10^{-6}$ $1.0\times10^{-6}$ | | Yaws (2003) Gharagheizi et al. (2012) Gharagheizi et al. (2010) | X Q Q | 237 246 |
| 4-methyl-5-ethyloctane $C_{11}H_{24}$ [62016-24-4] DTSHQAHKLUTGAR-UHFFFAOYSA-N | $1.1\times10^{-6}$ $1.2\times10^{-6}$ $1.0\times10^{-6}$ | | Yaws (2003) Gharagheizi et al. (2012) Gharagheizi et al. (2010) | X Q Q | 237 246 |
| 4-methyl-6-ethyloctane $C_{11}H_{24}$ [62016-25-5] NXUUVZOGBPCPDV-UHFFFAOYSA-N | $1.1\times10^{-6}$ $1.2\times10^{-6}$ $1.0\times10^{-6}$ | | Yaws (2003) Gharagheizi et al. (2012) Gharagheizi et al. (2010) | X Q Q | 237 246 |
| 4-propyloctane $C_{11}H_{24}$ [17302-13-5] VFAMBAFLNKONTN-UHFFFAOYSA-N | $1.1\times10^{-6}$ $1.2\times10^{-6}$ $1.2\times10^{-6}$ | | Yaws (2003) Gharagheizi et al. (2012) Gharagheizi et al. (2010) | X Q Q | 237 246 |
| 4-isopropyloctane $C_{11}H_{24}$ [62016-15-3] VSJAVEFYQMREHJ-UHFFFAOYSA-N | $1.1\times10^{-6}$ $1.1\times10^{-6}$ $1.0\times10^{-6}$ | | Yaws (2003) Gharagheizi et al. (2012) Gharagheizi et al. (2010) | X Q Q | 237 246 |
| 2,2,3,3-tetramethylheptane $C_{11}H_{24}$ [61868-40-4] YPGGGWSLWOECQF-UHFFFAOYSA-N | $7.5\times10^{-7}$ $1.2\times10^{-6}$ $8.1\times10^{-7}$ | | Yaws (2003) Gharagheizi et al. (2012) Gharagheizi et al. (2010) | X Q Q | 237 246 |
| 2,2,3,4-tetramethylheptane $C_{11}H_{24}$ [61868-41-5] HPKDRGPKZGWSND-UHFFFAOYSA-N | $8.6\times10^{-7}$ $1.1\times10^{-6}$ $8.7\times10^{-7}$ | | Yaws (2003) Gharagheizi et al. (2012) Gharagheizi et al. (2010) | X Q Q | 237 246 |
| 2,2,3,5-tetramethylheptane $C_{11}H_{24}$ [61868-42-6] KXNFNEAZTWCHIL-UHFFFAOYSA-N | $8.8\times10^{-7}$ $1.0\times10^{-6}$ $8.7\times10^{-7}$ | | Yaws (2003) Gharagheizi et al. (2012) Gharagheizi et al. (2010) | X Q Q | 237 246 |
| 2,2,3,6-tetramethylheptane $C_{11}H_{24}$ [61868-43-7] AQHGQPHAJYTMDJ-UHFFFAOYSA-N | $9.3\times10^{-7}$ $9.3\times10^{-7}$ $8.7\times10^{-7}$ | | Yaws (2003) Gharagheizi et al. (2012) Gharagheizi et al. (2010) | X Q Q | 237 246 |





Table A2.1: Alkanes (. . . continued)

| Substance<br>Formula<br>(Trivial Name)<br>[CAS Registry Number]<br>InChIKey | $H_s^{cp}$<br>(at $T^{\ominus}$)<br>$\left[\dfrac{\text{mol}}{\text{m}^3\,\text{Pa}}\right]$ | $\dfrac{\text{d}\ln H_s^{cp}}{\text{d}(1/T)}$<br><br>[K] | Reference | Type | Note |
|---|---|---|---|---|---|
| 2,2,4,4-tetramethylheptane<br>$C_{11}H_{24}$<br>[61868-44-8]<br>JMRHVFQTDSCACD-UHFFFAOYSA-N | $8.6\times10^{-7}$<br>$9.3\times10^{-7}$<br>$8.1\times10^{-7}$ | | Yaws (2003)<br>Gharagheizi et al. (2012)<br>Gharagheizi et al. (2010) | X<br>Q<br>Q | 237<br><br>246 |
| 2,2,4,5-tetramethylheptane<br>$C_{11}H_{24}$<br>[61868-45-9]<br>MPDBIZNQOUSUIN-UHFFFAOYSA-N | $9.5\times10^{-7}$<br>$9.0\times10^{-7}$<br>$8.7\times10^{-7}$ | | Yaws (2003)<br>Gharagheizi et al. (2012)<br>Gharagheizi et al. (2010) | X<br>Q<br>Q | 237<br><br>246 |
| 2,2,4,6-tetramethylheptane<br>$C_{11}H_{24}$<br>[61868-46-0]<br>XSMHBEAIQLABAO-UHFFFAOYSA-N | $1.0\times10^{-6}$<br>$7.4\times10^{-7}$<br>$8.7\times10^{-7}$ | | Yaws (2003)<br>Gharagheizi et al. (2012)<br>Gharagheizi et al. (2010) | X<br>Q<br>Q | 237<br><br>246 |
| 2,2,5,5-tetramethylheptane<br>$C_{11}H_{24}$<br>[61868-47-1]<br>VQJNSRPWQMPDDO-UHFFFAOYSA-N | $9.1\times10^{-7}$<br>$8.3\times10^{-7}$<br>$8.1\times10^{-7}$ | | Yaws (2003)<br>Gharagheizi et al. (2012)<br>Gharagheizi et al. (2010) | X<br>Q<br>Q | 237<br><br>246 |
| 2,2,5,6-tetramethylheptane<br>$C_{11}H_{24}$<br>[61868-48-2]<br>ZYJXKOJVCNGFSL-UHFFFAOYSA-N | $9.5\times10^{-7}$<br>$9.0\times10^{-7}$<br>$8.7\times10^{-7}$ | | Yaws (2003)<br>Gharagheizi et al. (2012)<br>Gharagheizi et al. (2010) | X<br>Q<br>Q | 237<br><br>246 |
| 2,2,6,6-tetramethylheptane<br>$C_{11}H_{24}$<br>[40117-45-1]<br>GKNMBVVJQTWDRT-UHFFFAOYSA-N | $9.5\times10^{-7}$<br>$7.6\times10^{-7}$<br>$8.1\times10^{-7}$ | | Yaws (2003)<br>Gharagheizi et al. (2012)<br>Gharagheizi et al. (2010) | X<br>Q<br>Q | 237<br><br>246 |
| 2,3,3,4-tetramethylheptane<br>$C_{11}H_{24}$<br>[61868-49-3]<br>OHIUILFTBGSFAG-UHFFFAOYSA-N | $8.0\times10^{-7}$<br>$1.3\times10^{-6}$<br>$8.7\times10^{-7}$ | | Yaws (2003)<br>Gharagheizi et al. (2012)<br>Gharagheizi et al. (2010) | X<br>Q<br>Q | 237<br><br>246 |
| 2,3,3,5-tetramethylheptane<br>$C_{11}H_{24}$<br>[61868-50-6]<br>XGCFPCZVDOZHLT-UHFFFAOYSA-N | $8.6\times10^{-7}$<br>$1.1\times10^{-6}$<br>$8.7\times10^{-7}$ | | Yaws (2003)<br>Gharagheizi et al. (2012)<br>Gharagheizi et al. (2010) | X<br>Q<br>Q | 237<br><br>246 |
| 2,3,3,6-tetramethylheptane<br>$C_{11}H_{24}$<br>[61868-51-7]<br>ZVRALCAHDAJURU-UHFFFAOYSA-N | $8.7\times10^{-7}$<br>$1.1\times10^{-6}$<br>$8.7\times10^{-7}$ | | Yaws (2003)<br>Gharagheizi et al. (2012)<br>Gharagheizi et al. (2010) | X<br>Q<br>Q | 237<br><br>246 |
| 2,3,4,4-tetramethylheptane<br>$C_{11}H_{24}$<br>[61868-52-8]<br>SXHQWTUVNZIMTP-UHFFFAOYSA-N | $8.4\times10^{-7}$<br>$1.2\times10^{-6}$<br>$8.7\times10^{-7}$ | | Yaws (2003)<br>Gharagheizi et al. (2012)<br>Gharagheizi et al. (2010) | X<br>Q<br>Q | 237<br><br>246 |





Table A2.1: Alkanes (... continued)

| Substance Formula (Trivial Name) [CAS Registry Number] InChIKey | $H_s^{cp}$ (at $T^\ominus$) $\left[\dfrac{\mathrm{mol}}{\mathrm{m}^3\,\mathrm{Pa}}\right]$ | $\dfrac{\mathrm{d}\ln H_s^{cp}}{\mathrm{d}(1/T)}$ [K] | Reference | Type | Note |
|---|---|---|---|---|---|
| 2,3,4,5-tetramethylheptane $C_{11}H_{24}$ [61868-53-9] LGUFKOOYNDNVNP-UHFFFAOYSA-N | $9.1\times10^{-7}$ $1.2\times10^{-6}$ $9.6\times10^{-7}$ | | Yaws (2003) Gharagheizi et al. (2012) Gharagheizi et al. (2010) | X Q Q | 237 246 |
| 2,3,4,6-tetramethylheptane $C_{11}H_{24}$ [61868-54-0] WOHJQOFSYZPITE-UHFFFAOYSA-N | $9.6\times10^{-7}$ $1.0\times10^{-6}$ $9.6\times10^{-7}$ | | Yaws (2003) Gharagheizi et al. (2012) Gharagheizi et al. (2010) | X Q Q | 237 246 |
| 2,3,5,5-tetramethylheptane $C_{11}H_{24}$ [61868-55-1] QSBIXVGCKCYBBL-UHFFFAOYSA-N | $9.1\times10^{-7}$ $9.8\times10^{-7}$ $8.7\times10^{-7}$ | | Yaws (2003) Gharagheizi et al. (2012) Gharagheizi et al. (2010) | X Q Q | 237 246 |
| 2,3,5,6-tetramethylheptane $C_{11}H_{24}$ [52670-32-3] RAHGVMDMAJFLTP-UHFFFAOYSA-N | $9.5\times10^{-7}$ $1.1\times10^{-6}$ $9.6\times10^{-7}$ | | Yaws (2003) Gharagheizi et al. (2012) Gharagheizi et al. (2010) | X Q Q | 237 246 |
| 2,4,4,5-tetramethylheptane $C_{11}H_{24}$ [61868-56-2] QALMSCCHYDFGLF-UHFFFAOYSA-N | $8.8\times10^{-7}$ $1.0\times10^{-6}$ $8.7\times10^{-7}$ | | Yaws (2003) Gharagheizi et al. (2012) Gharagheizi et al. (2010) | X Q Q | 237 246 |
| 2,4,4,6-tetramethylheptane $C_{11}H_{24}$ [61868-57-3] DTMSZIVVBLZFEV-UHFFFAOYSA-N | $1.0\times10^{-6}$ $7.8\times10^{-7}$ $8.7\times10^{-7}$ | | Yaws (2003) Gharagheizi et al. (2012) Gharagheizi et al. (2010) | X Q Q | 237 246 |
| 2,4,5,5-tetramethylheptane $C_{11}H_{24}$ [61868-58-4] JVYVKEOAYQAABU-UHFFFAOYSA-N | $8.6\times10^{-7}$ $1.1\times10^{-6}$ $8.7\times10^{-7}$ | | Yaws (2003) Gharagheizi et al. (2012) Gharagheizi et al. (2010) | X Q Q | 237 246 |
| 3,3,4,4-tetramethylheptane $C_{11}H_{24}$ [61868-59-5] CFNIIBCZVLNKPV-UHFFFAOYSA-N | $7.2\times10^{-7}$ $1.4\times10^{-6}$ $8.1\times10^{-7}$ | | Yaws (2003) Gharagheizi et al. (2012) Gharagheizi et al. (2010) | X Q Q | 237 246 |
| 3,3,4,5-tetramethylheptane $C_{11}H_{24}$ [61868-60-8] QVCSKDHUHWRVTA-UHFFFAOYSA-N | $8.1\times10^{-7}$ $1.3\times10^{-6}$ $8.7\times10^{-7}$ | | Yaws (2003) Gharagheizi et al. (2012) Gharagheizi et al. (2010) | X Q Q | 237 246 |
| 3,3,5,5-tetramethylheptane $C_{11}H_{24}$ [61868-61-9] GALDMESQGNRJGE-UHFFFAOYSA-N | $7.5\times10^{-7}$ $1.2\times10^{-6}$ $8.1\times10^{-7}$ | | Yaws (2003) Gharagheizi et al. (2012) Gharagheizi et al. (2010) | X Q Q | 237 246 |





Table A2.1: Alkanes (...continued)

| Substance Formula (Trivial Name) [CAS Registry Number] InChIKey | $H_s^{cp}$ (at $T^{\ominus}$) $\left[\dfrac{\mathrm{mol}}{\mathrm{m^3\,Pa}}\right]$ | $\dfrac{\mathrm{d}\ln H_s^{cp}}{\mathrm{d}(1/T)}$ [K] | Reference | Type | Note |
|---|---|---|---|---|---|
| 3,4,4,5-tetramethylheptane C$_{11}$H$_{24}$ [61868-62-0] CXIAIMLBUJNNJR-UHFFFAOYSA-N | $7.8\times10^{-7}$ $1.4\times10^{-6}$ $8.7\times10^{-7}$ | | Yaws (2003) Gharagheizi et al. (2012) Gharagheizi et al. (2010) | X Q Q | 237 246 |
| 2,2-dimethyl-3-ethylheptane C$_{11}$H$_{24}$ [61869-03-2] SBWMIVQVGGUMCE-UHFFFAOYSA-N | $9.3\times10^{-7}$ $1.1\times10^{-6}$ $9.0\times10^{-7}$ | | Yaws (2003) Gharagheizi et al. (2012) Gharagheizi et al. (2010) | X Q Q | 237 246 |
| 2,2-dimethyl-4-ethylheptane C$_{11}$H$_{24}$ [62016-46-0] BJOUMNMMNSZESV-UHFFFAOYSA-N | $1.0\times10^{-6}$ $8.7\times10^{-7}$ $9.0\times10^{-7}$ | | Yaws (2003) Gharagheizi et al. (2012) Gharagheizi et al. (2010) | X Q Q | 237 246 |
| 2,2-dimethyl-5-ethylheptane C$_{11}$H$_{24}$ [62016-47-1] HVVFIKMZXFNXKA-UHFFFAOYSA-N | $9.8\times10^{-7}$ $9.7\times10^{-7}$ $9.0\times10^{-7}$ | | Yaws (2003) Gharagheizi et al. (2012) Gharagheizi et al. (2010) | X Q Q | 237 246 |
| 2,3-dimethyl-3-ethylheptane C$_{11}$H$_{24}$ [61868-21-1] KQUHXOYVQZYOOF-UHFFFAOYSA-N | $8.9\times10^{-7}$ $1.2\times10^{-6}$ $9.0\times10^{-7}$ | | Yaws (2003) Gharagheizi et al. (2012) Gharagheizi et al. (2010) | X Q Q | 237 246 |
| 2,3-dimethyl-4-ethylheptane C$_{11}$H$_{24}$ [61868-22-2] CNEPAGDSOHLVRW-UHFFFAOYSA-N | $9.8\times10^{-7}$ $1.2\times10^{-6}$ $9.7\times10^{-7}$ | | Yaws (2003) Gharagheizi et al. (2012) Gharagheizi et al. (2010) | X Q Q | 237 246 |
| 2,3-dimethyl-5-ethylheptane C$_{11}$H$_{24}$ [61868-23-3] BFCKNSGKYWMUNP-UHFFFAOYSA-N | $1.0\times10^{-6}$ $1.2\times10^{-6}$ $9.7\times10^{-7}$ | | Yaws (2003) Gharagheizi et al. (2012) Gharagheizi et al. (2010) | X Q Q | 237 246 |
| 2,4-dimethyl-3-ethylheptane C$_{11}$H$_{24}$ [61868-24-4] OBCGEYFMJYXFJV-UHFFFAOYSA-N | $9.8\times10^{-7}$ $1.2\times10^{-6}$ $9.7\times10^{-7}$ | | Yaws (2003) Gharagheizi et al. (2012) Gharagheizi et al. (2010) | X Q Q | 237 246 |
| 2,4-dimethyl-4-ethylheptane C$_{11}$H$_{24}$ [61868-25-5] OETKSNKIFXLJDO-UHFFFAOYSA-N | $1.0\times10^{-6}$ $9.2\times10^{-7}$ $9.0\times10^{-7}$ | | Yaws (2003) Gharagheizi et al. (2012) Gharagheizi et al. (2010) | X Q Q | 237 246 |
| 2,4-dimethyl-5-ethylheptane C$_{11}$H$_{24}$ [61868-26-6] RZHNOEUMKYRHIX-UHFFFAOYSA-N | $1.0\times10^{-6}$ $1.1\times10^{-6}$ $9.7\times10^{-7}$ | | Yaws (2003) Gharagheizi et al. (2012) Gharagheizi et al. (2010) | X Q Q | 237 246 |



Table A2.1: Alkanes (. . . continued)

| Substance<br>Formula<br>(Trivial Name)<br>[CAS Registry Number]<br>InChIKey | $H_s^{cp}$<br>(at $T^\ominus$)<br><br>$\left[\dfrac{\text{mol}}{\text{m}^3\,\text{Pa}}\right]$ | $\dfrac{\text{d}\ln H_s^{cp}}{\text{d}(1/T)}$<br><br>[K] | Reference | Type | Note |
|---|---|---|---|---|---|
| 2,5-dimethyl-3-ethylheptane<br>$C_{11}H_{24}$<br>[61868-27-7]<br>DEZOXTLNRYCKNK-UHFFFAOYSA-N | $1.0\times10^{-6}$<br>$1.1\times10^{-6}$<br>$9.7\times10^{-7}$ | | Yaws (2003)<br>Gharagheizi et al. (2012)<br>Gharagheizi et al. (2010) | X<br>Q<br>Q | 237<br><br>246 |
| 2,5-dimethyl-4-ethylheptane<br>$C_{11}H_{24}$<br>[61868-28-8]<br>CUHCBWKQRDWQMN-UHFFFAOYSA-N | $1.0\times10^{-6}$<br>$1.1\times10^{-6}$<br>$9.7\times10^{-7}$ | | Yaws (2003)<br>Gharagheizi et al. (2012)<br>Gharagheizi et al. (2010) | X<br>Q<br>Q | 237<br><br>246 |
| 2,5-dimethyl-5-ethylheptane<br>$C_{11}H_{24}$<br>[61868-29-9]<br>GPZDEXDVXFJVLH-UHFFFAOYSA-N | $9.1\times10^{-7}$<br>$1.2\times10^{-6}$<br>$9.0\times10^{-7}$ | | Yaws (2003)<br>Gharagheizi et al. (2012)<br>Gharagheizi et al. (2010) | X<br>Q<br>Q | 237<br><br>246 |
| 2,6-dimethyl-3-ethylheptane<br>$C_{11}H_{24}$<br>[61868-30-2]<br>BFXHSWDHLPKFSH-UHFFFAOYSA-N | $1.0\times10^{-6}$<br>$1.1\times10^{-6}$<br>$9.7\times10^{-7}$ | | Yaws (2003)<br>Gharagheizi et al. (2012)<br>Gharagheizi et al. (2010) | X<br>Q<br>Q | 237<br><br>246 |
| 2,6-dimethyl-4-ethylheptane<br>$C_{11}H_{24}$<br>[61868-31-3]<br>GBSGJLJGXHBTQQ-UHFFFAOYSA-N | $1.1\times10^{-6}$<br>$9.2\times10^{-7}$<br>$9.7\times10^{-7}$ | | Yaws (2003)<br>Gharagheizi et al. (2012)<br>Gharagheizi et al. (2010) | X<br>Q<br>Q | 237<br><br>246 |
| 3,3-dimethyl-4-ethylheptane<br>$C_{11}H_{24}$<br>[61868-32-4]<br>QXJXAHFNWHEQEU-UHFFFAOYSA-N | $8.9\times10^{-7}$<br>$1.2\times10^{-6}$<br>$9.0\times10^{-7}$ | | Yaws (2003)<br>Gharagheizi et al. (2012)<br>Gharagheizi et al. (2010) | X<br>Q<br>Q | 237<br><br>246 |
| 3,3-dimethyl-5-ethylheptane<br>$C_{11}H_{24}$<br>[61868-33-5]<br>FLLHBJKGZIOYAA-UHFFFAOYSA-N | $9.4\times10^{-7}$<br>$1.1\times10^{-6}$<br>$9.0\times10^{-7}$ | | Yaws (2003)<br>Gharagheizi et al. (2012)<br>Gharagheizi et al. (2010) | X<br>Q<br>Q | 237<br><br>246 |
| 3,4-dimethyl-3-ethylheptane<br>$C_{11}H_{24}$<br>[61868-34-6]<br>HKSCZRRMBTUZLO-UHFFFAOYSA-N | $8.5\times10^{-7}$<br>$1.4\times10^{-6}$<br>$9.0\times10^{-7}$ | | Yaws (2003)<br>Gharagheizi et al. (2012)<br>Gharagheizi et al. (2010) | X<br>Q<br>Q | 237<br><br>246 |
| 3,4-dimethyl-4-ethylheptane<br>$C_{11}H_{24}$<br>[61868-35-7]<br>RMVWUAZCGGWZKA-UHFFFAOYSA-N | $8.8\times10^{-7}$<br>$1.3\times10^{-6}$<br>$9.0\times10^{-7}$ | | Yaws (2003)<br>Gharagheizi et al. (2012)<br>Gharagheizi et al. (2010) | X<br>Q<br>Q | 237<br><br>246 |
| 3,4-dimethyl-5-ethylheptane<br>$C_{11}H_{24}$<br>[61868-36-8]<br>PCRCVFLPWIQGNJ-UHFFFAOYSA-N | $9.5\times10^{-7}$<br>$1.3\times10^{-6}$<br>$9.7\times10^{-7}$ | | Yaws (2003)<br>Gharagheizi et al. (2012)<br>Gharagheizi et al. (2010) | X<br>Q<br>Q | 237<br><br>246 |



Table A2.1: Alkanes (...continued)

| Substance Formula (Trivial Name) [CAS Registry Number] InChIKey | $H_s^{cp}$ (at $T^\ominus$) $\left[\dfrac{\mathrm{mol}}{\mathrm{m}^3\,\mathrm{Pa}}\right]$ | $\dfrac{\mathrm{d}\ln H_s^{cp}}{\mathrm{d}(1/T)}$ [K] | Reference | Type | Note |
|---|---|---|---|---|---|
| 3,5-dimethyl-3-ethylheptane $C_{11}H_{24}$ [61868-37-9] FQOCKNRDCVODKW-UHFFFAOYSA-N | $8.9\times10^{-7}$ $1.2\times10^{-6}$ $9.0\times10^{-7}$ | | Yaws (2003) Gharagheizi et al. (2012) Gharagheizi et al. (2010) | X Q Q | 237 246 |
| 3,5-dimethyl-4-ethylheptane $C_{11}H_{24}$ [61868-38-0] IKVNPWSSYMJDKE-UHFFFAOYSA-N | $9.6\times10^{-7}$ $1.3\times10^{-6}$ $9.7\times10^{-7}$ | | Yaws (2003) Gharagheizi et al. (2012) Gharagheizi et al. (2010) | X Q Q | 237 246 |
| 4,4-dimethyl-3-ethylheptane $C_{11}H_{24}$ [61868-39-1] YMWPSGKMCMMFSS-UHFFFAOYSA-N | $9.3\times10^{-7}$ $1.1\times10^{-6}$ $9.0\times10^{-7}$ | | Yaws (2003) Gharagheizi et al. (2012) Gharagheizi et al. (2010) | X Q Q | 237 246 |
| 2-methyl-4-propylheptane $C_{11}H_{24}$ [61868-96-0] AXOUMHTUGUFOKF-UHFFFAOYSA-N | $1.1\times10^{-6}$ $1.0\times10^{-6}$ $1.0\times10^{-6}$ | | Yaws (2003) Gharagheizi et al. (2012) Gharagheizi et al. (2010) | X Q Q | 237 246 |
| 3-methyl-4-propylheptane $C_{11}H_{24}$ [61868-97-1] WAMROSASDOIVDE-UHFFFAOYSA-N | $1.1\times10^{-6}$ $1.2\times10^{-6}$ $1.0\times10^{-6}$ | | Yaws (2003) Gharagheizi et al. (2012) Gharagheizi et al. (2010) | X Q Q | 237 246 |
| 4-methyl-4-propylheptane $C_{11}H_{24}$ [17302-20-4] RDZFGBZTNOYNAH-UHFFFAOYSA-N | $9.9\times10^{-7}$ $1.1\times10^{-6}$ $1.0\times10^{-6}$ | | Yaws (2003) Gharagheizi et al. (2012) Gharagheizi et al. (2010) | X Q Q | 237 246 |
| 2-methyl-3-isopropylheptane $C_{11}H_{24}$ [6876-18-2] GECZBVJYUAPYFD-UHFFFAOYSA-N | $9.9\times10^{-7}$ $1.2\times10^{-6}$ $9.7\times10^{-7}$ | | Yaws (2003) Gharagheizi et al. (2012) Gharagheizi et al. (2010) | X Q Q | 237 246 |
| 2-methyl-4-isopropylheptane $C_{11}H_{24}$ [61868-98-2] ADLZFTVQXMDTCY-UHFFFAOYSA-N | $1.1\times10^{-6}$ $9.7\times10^{-7}$ $9.7\times10^{-7}$ | | Yaws (2003) Gharagheizi et al. (2012) Gharagheizi et al. (2010) | X Q Q | 237 246 |
| 3-methyl-4-isopropylheptane $C_{11}H_{24}$ [61868-99-3] SRECVVXQFXRBNI-UHFFFAOYSA-N | $1.0\times10^{-6}$ $1.1\times10^{-6}$ $9.7\times10^{-7}$ | | Yaws (2003) Gharagheizi et al. (2012) Gharagheizi et al. (2010) | X Q Q | 237 246 |
| 4-methyl-4-isopropylheptane $C_{11}H_{24}$ [61869-00-9] LSJZMBVEVGYTAG-UHFFFAOYSA-N | $8.9\times10^{-7}$ $1.2\times10^{-6}$ $9.0\times10^{-7}$ | | Yaws (2003) Gharagheizi et al. (2012) Gharagheizi et al. (2010) | X Q Q | 237 246 |





Table A2.1: Alkanes (. . . continued)

| Substance<br>Formula<br>(Trivial Name)<br>[CAS Registry Number]<br>InChIKey | $H_s^{cp}$<br>(at $T^\ominus$)<br>$\left[\dfrac{\text{mol}}{\text{m}^3\,\text{Pa}}\right]$ | $\dfrac{\text{d}\ln H_s^{cp}}{\text{d}(1/T)}$<br><br>[K] | Reference | Type | Note |
|---|---|---|---|---|---|
| 3,3-diethylheptane<br>$C_{11}H_{24}$<br>[17302-17-9]<br>XATIZWAWQAIMQJ-UHFFFAOYSA-N | $9.1\times10^{-7}$<br>$1.5\times10^{-6}$<br>$1.0\times10^{-6}$ | | Yaws (2003)<br>Gharagheizi et al. (2012)<br>Gharagheizi et al. (2010) | X<br>Q<br>Q | 237<br><br>246 |
| 3,4-diethylheptane<br>$C_{11}H_{24}$<br>[61869-01-0]<br>UWOPVDFSBGDRJR-UHFFFAOYSA-N | $1.1\times10^{-6}$<br>$1.2\times10^{-6}$<br>$1.0\times10^{-6}$ | | Yaws (2003)<br>Gharagheizi et al. (2012)<br>Gharagheizi et al. (2010) | X<br>Q<br>Q | 237<br><br>246 |
| 3,5-diethylheptane<br>$C_{11}H_{24}$<br>[61869-02-1]<br>OBHTWTQZINRNAB-UHFFFAOYSA-N | $1.1\times10^{-6}$<br>$1.2\times10^{-6}$<br>$1.0\times10^{-6}$ | | Yaws (2003)<br>Gharagheizi et al. (2012)<br>Gharagheizi et al. (2010) | X<br>Q<br>Q | 237<br><br>246 |
| 4,4-diethylheptane<br>$C_{11}H_{24}$<br>[17302-21-5]<br>WDTMGYSKSAMSPF-UHFFFAOYSA-N | $9.6\times10^{-7}$<br>$1.2\times10^{-6}$<br>$1.0\times10^{-6}$ | | Yaws (2003)<br>Gharagheizi et al. (2012)<br>Gharagheizi et al. (2010) | X<br>Q<br>Q | 237<br><br>246 |
| 4-*tert*-butylheptane<br>$C_{11}H_{24}$<br>[60302-21-8]<br>MDOHZJRFNISIIH-UHFFFAOYSA-N | $9.6\times10^{-7}$<br>$1.0\times10^{-6}$<br>$9.0\times10^{-7}$ | | Yaws (2003)<br>Gharagheizi et al. (2012)<br>Gharagheizi et al. (2010) | X<br>Q<br>Q | 237<br><br>246 |
| 2,2,3,3,4-pentamethylhexane<br>$C_{11}H_{24}$<br>[61868-85-7]<br>RSEBTOONCGVAKN-UHFFFAOYSA-N | $6.6\times10^{-7}$<br>$1.4\times10^{-6}$<br>$8.1\times10^{-7}$ | | Yaws (2003)<br>Gharagheizi et al. (2012)<br>Gharagheizi et al. (2010) | X<br>Q<br>Q | 237<br><br>246 |
| 2,2,3,3,5-pentamethylhexane<br>$C_{11}H_{24}$<br>[61868-86-8]<br>QGVXFIIZUSICQO-UHFFFAOYSA-N | $7.9\times10^{-7}$<br>$9.5\times10^{-7}$<br>$8.1\times10^{-7}$ | | Yaws (2003)<br>Gharagheizi et al. (2012)<br>Gharagheizi et al. (2010) | X<br>Q<br>Q | 237<br><br>246 |
| 2,2,3,4,4-pentamethylhexane<br>$C_{11}H_{24}$<br>[61868-87-9]<br>HBVQLCKOFAUQGY-UHFFFAOYSA-N | $6.5\times10^{-7}$<br>$1.5\times10^{-6}$<br>$8.1\times10^{-7}$ | | Yaws (2003)<br>Gharagheizi et al. (2012)<br>Gharagheizi et al. (2010) | X<br>Q<br>Q | 237<br><br>246 |
| 2,2,3,4,5-pentamethylhexane<br>$C_{11}H_{24}$<br>[61868-88-0]<br>LDUJDGROUWHEKU-UHFFFAOYSA-N | $8.2\times10^{-7}$<br>$1.1\times10^{-6}$<br>$9.0\times10^{-7}$ | | Yaws (2003)<br>Gharagheizi et al. (2012)<br>Gharagheizi et al. (2010) | X<br>Q<br>Q | 237<br><br>246 |
| 2,2,3,5,5-pentamethylhexane<br>$C_{11}H_{24}$<br>[14739-73-2]<br>UKYAYXXQXKPIBL-UHFFFAOYSA-N | $8.5\times10^{-7}$<br>$8.4\times10^{-7}$<br>$8.1\times10^{-7}$ | | Yaws (2003)<br>Gharagheizi et al. (2012)<br>Gharagheizi et al. (2010) | X<br>Q<br>Q | 237<br><br>246 |





Table A2.1: Alkanes (. . . continued)

| Substance Formula (Trivial Name) [CAS Registry Number] InChIKey | $H_s^{cp}$ (at $T^{\ominus}$) $\left[\dfrac{\mathrm{mol}}{\mathrm{m}^3\,\mathrm{Pa}}\right]$ | $\dfrac{\mathrm{d}\ln H_s^{cp}}{\mathrm{d}(1/T)}$ [K] | Reference | Type | Note |
|---|---|---|---|---|---|
| 2,2,4,4,5-pentamethylhexane $C_{11}H_{24}$ [60302-23-0] JQFZWBZEZYOCQL-UHFFFAOYSA-N | $7.3\times10^{-7}$ $1.1\times10^{-6}$ $8.1\times10^{-7}$ | | Yaws (2003) Gharagheizi et al. (2012) Gharagheizi et al. (2010) | X Q Q | 237 246 |
| 2,3,3,4,4-pentamethylhexane $C_{11}H_{24}$ [61868-89-1] GHRPCKCGHFDDOH-UHFFFAOYSA-N | $6.4\times10^{-7}$ $1.5\times10^{-6}$ $8.1\times10^{-7}$ | | Yaws (2003) Gharagheizi et al. (2012) Gharagheizi et al. (2010) | X Q Q | 237 246 |
| 2,3,3,4,5-pentamethylhexane $C_{11}H_{24}$ [52670-33-4] STMIMOZUYGUOQZ-UHFFFAOYSA-N | $7.5\times10^{-7}$ $1.3\times10^{-6}$ $9.0\times10^{-7}$ | | Yaws (2003) Gharagheizi et al. (2012) Gharagheizi et al. (2010) | X Q Q | 237 246 |
| 2,2,3-trimethyl-3-ethylhexane $C_{11}H_{24}$ [61868-72-2] HUAYVWMWWLHZAB-UHFFFAOYSA-N | $7.5\times10^{-7}$ $1.2\times10^{-6}$ $8.1\times10^{-7}$ | | Yaws (2003) Gharagheizi et al. (2012) Gharagheizi et al. (2010) | X Q Q | 237 246 |
| 2,2,3-trimethyl-4-ethylhexane $C_{11}H_{24}$ [61868-73-3] ZSUMTZIXJNZKBI-UHFFFAOYSA-N | $8.6\times10^{-7}$ $1.1\times10^{-6}$ $8.7\times10^{-7}$ | | Yaws (2003) Gharagheizi et al. (2012) Gharagheizi et al. (2010) | X Q Q | 237 246 |
| 2,2,4-trimethyl-3-ethylhexane $C_{11}H_{24}$ [61868-74-4] UDUTWNYTAVENNO-UHFFFAOYSA-N | $8.7\times10^{-7}$ $1.1\times10^{-6}$ $8.7\times10^{-7}$ | | Yaws (2003) Gharagheizi et al. (2012) Gharagheizi et al. (2010) | X Q Q | 237 246 |
| 2,2,4-trimethyl-4-ethylhexane $C_{11}H_{24}$ [61868-75-5] YFQDCXGMLYJDSG-UHFFFAOYSA-N | $7.7\times10^{-7}$ $1.2\times10^{-6}$ $8.1\times10^{-7}$ | | Yaws (2003) Gharagheizi et al. (2012) Gharagheizi et al. (2010) | X Q Q | 237 246 |
| 2,2,5-trimethyl-3-ethylhexane $C_{11}H_{24}$ [61868-76-6] GJVFOLLEVKEKMU-UHFFFAOYSA-N | $9.3\times10^{-7}$ $9.3\times10^{-7}$ $8.7\times10^{-7}$ | | Yaws (2003) Gharagheizi et al. (2012) Gharagheizi et al. (2010) | X Q Q | 237 246 |
| 2,2,5-trimethyl-4-ethylhexane $C_{11}H_{24}$ [61868-77-7] UKTBERWYLDSZDQ-UHFFFAOYSA-N | $9.9\times10^{-7}$ $8.3\times10^{-7}$ $8.7\times10^{-7}$ | | Yaws (2003) Gharagheizi et al. (2012) Gharagheizi et al. (2010) | X Q Q | 237 246 |
| 2,3,3-trimethyl-4-ethylhexane $C_{11}H_{24}$ [61868-78-8] DNDCULLMOLSZER-UHFFFAOYSA-N | $7.9\times10^{-7}$ $1.4\times10^{-6}$ $8.7\times10^{-7}$ | | Yaws (2003) Gharagheizi et al. (2012) Gharagheizi et al. (2010) | X Q Q | 237 246 |





Table A2.1: Alkanes (... continued)

| Substance Formula (Trivial Name) [CAS Registry Number] InChIKey | $H_s^{cp}$ (at $T^\ominus$) $\left[\dfrac{\text{mol}}{\text{m}^3\,\text{Pa}}\right]$ | $\dfrac{\mathrm{d}\ln H_s^{cp}}{\mathrm{d}(1/T)}$ [K] | Reference | Type | Note |
|---|---|---|---|---|---|
| 2,3,4-trimethyl-3-ethylhexane $C_{11}H_{24}$ [61868-79-9] XCNOKZUAQAZTAJ-UHFFFAOYSA-N | $7.6\times10^{-7}$ $1.5\times10^{-6}$ $8.7\times10^{-7}$ | | Yaws (2003) Gharagheizi et al. (2012) Gharagheizi et al. (2010) | X Q Q | 237 246 |
| 2,3,4-trimethyl-4-ethylhexane $C_{11}H_{24}$ [61868-80-2] HNNSQDJQJRTMMW-UHFFFAOYSA-N | $8.2\times10^{-7}$ $1.2\times10^{-6}$ $8.7\times10^{-7}$ | | Yaws (2003) Gharagheizi et al. (2012) Gharagheizi et al. (2010) | X Q Q | 237 246 |
| 2,3,5-trimethyl-3-ethylhexane $C_{11}H_{24}$ [61868-81-3] VYVPEEFLIYVOTH-UHFFFAOYSA-N | $8.6\times10^{-7}$ $1.1\times10^{-6}$ $8.7\times10^{-7}$ | | Yaws (2003) Gharagheizi et al. (2012) Gharagheizi et al. (2010) | X Q Q | 237 246 |
| 2,3,5-trimethyl-4-ethylhexane $C_{11}H_{24}$ [61868-82-4] LDKMZLRIVWGQRW-UHFFFAOYSA-N | $9.3\times10^{-7}$ $1.1\times10^{-6}$ $9.6\times10^{-7}$ | | Yaws (2003) Gharagheizi et al. (2012) Gharagheizi et al. (2010) | X Q Q | 237 246 |
| 2,4,4-trimethyl-3-ethylhexane $C_{11}H_{24}$ [61868-83-5] MTYPKDRKJRWXKB-UHFFFAOYSA-N | $8.4\times10^{-7}$ $1.2\times10^{-6}$ $8.7\times10^{-7}$ | | Yaws (2003) Gharagheizi et al. (2012) Gharagheizi et al. (2010) | X Q Q | 237 246 |
| 3,3,4-trimethyl-4-ethylhexane $C_{11}H_{24}$ [61868-84-6] ADDYWUNUHVKQGT-UHFFFAOYSA-N | $6.8\times10^{-7}$ $1.7\times10^{-6}$ $8.1\times10^{-7}$ | | Yaws (2003) Gharagheizi et al. (2012) Gharagheizi et al. (2010) | X Q Q | 237 246 |
| 2,2-dimethyl-3-isopropylhexane $C_{11}H_{24}$ [61868-63-1] VKABOTNMFKIVDN-UHFFFAOYSA-N | $9.1\times10^{-7}$ $9.8\times10^{-7}$ $8.7\times10^{-7}$ | | Yaws (2003) Gharagheizi et al. (2012) Gharagheizi et al. (2010) | X Q Q | 237 246 |
| 2,3-dimethyl-3-isopropylhexane $C_{11}H_{24}$ [61868-64-2] XVWGUCNHEXRNJU-UHFFFAOYSA-N | $8.2\times10^{-7}$ $1.2\times10^{-6}$ $8.7\times10^{-7}$ | | Yaws (2003) Gharagheizi et al. (2012) Gharagheizi et al. (2010) | X Q Q | 237 246 |
| 2,4-dimethyl-3-isopropylhexane $C_{11}H_{24}$ [61868-65-3] QRNSQXWPYPLHTL-UHFFFAOYSA-N | $9.5\times10^{-7}$ $1.1\times10^{-6}$ $9.6\times10^{-7}$ | | Yaws (2003) Gharagheizi et al. (2012) Gharagheizi et al. (2010) | X Q Q | 237 246 |
| 2,5-dimethyl-3-isopropylhexane $C_{11}H_{24}$ [61868-66-4] QKICRGDGSLQROM-UHFFFAOYSA-N | $1.0\times10^{-6}$ $9.2\times10^{-7}$ $9.6\times10^{-7}$ | | Yaws (2003) Gharagheizi et al. (2012) Gharagheizi et al. (2010) | X Q Q | 237 246 |



Table A2.1: Alkanes (...continued)

| Substance Formula (Trivial Name) [CAS Registry Number] InChIKey | $H_s^{cp}$ (at $T^\ominus$) $\left[\dfrac{\mathrm{mol}}{\mathrm{m^3\,Pa}}\right]$ | $\dfrac{\mathrm{d}\ln H_s^{cp}}{\mathrm{d}(1/T)}$ [K] | Reference | Type | Note |
|---|---|---|---|---|---|
| 2-methyl-3,3-diethylhexane $C_{11}H_{24}$ [61868-67-5] QNRHKXCWHYUEKP-UHFFFAOYSA-N | $8.4\times10^{-7}$ $1.5\times10^{-6}$ $9.0\times10^{-7}$ | | Yaws (2003) Gharagheizi et al. (2012) Gharagheizi et al. (2010) | X Q Q | 237 246 |
| 2-methyl-3,4-diethylhexane $C_{11}H_{24}$ [61868-68-6] MWHIPTVSAPEYPC-UHFFFAOYSA-N | $9.8\times10^{-7}$ $1.2\times10^{-6}$ $9.7\times10^{-7}$ | | Yaws (2003) Gharagheizi et al. (2012) Gharagheizi et al. (2010) | X Q Q | 237 246 |
| 2-methyl-4,4-diethylhexane $C_{11}H_{24}$ [61868-69-7] UNPXTJSCIHOMKB-UHFFFAOYSA-N | $9.0\times10^{-7}$ $1.2\times10^{-6}$ $9.0\times10^{-7}$ | | Yaws (2003) Gharagheizi et al. (2012) Gharagheizi et al. (2010) | X Q Q | 237 246 |
| 3-methyl-3,4-diethylhexane $C_{11}H_{24}$ [61868-70-0] ZFMWIWWLOKHXHC-UHFFFAOYSA-N | $8.4\times10^{-7}$ $1.5\times10^{-6}$ $9.0\times10^{-7}$ | | Yaws (2003) Gharagheizi et al. (2012) Gharagheizi et al. (2010) | X Q Q | 237 246 |
| 3-methyl-4,4-diethylhexane $C_{11}H_{24}$ [61868-71-1] CUHCCZRBXHOAMW-UHFFFAOYSA-N | $8.1\times10^{-7}$ $1.6\times10^{-6}$ $9.0\times10^{-7}$ | | Yaws (2003) Gharagheizi et al. (2012) Gharagheizi et al. (2010) | X Q Q | 237 246 |
| 2,2,3,3,4,4-hexamethylpentane $C_{11}H_{24}$ [60302-27-4] JKJQSSSRSKPVEM-UHFFFAOYSA-N | $5.0\times10^{-7}$ $1.9\times10^{-6}$ $7.9\times10^{-7}$ | | Yaws (2003) Gharagheizi et al. (2012) Gharagheizi et al. (2010) | X Q Q | 237 246 |
| 2,2,3,4-tetramethyl-3-ethylpentane $C_{11}H_{24}$ [61868-93-7] HBSINJYUKWVYBC-UHFFFAOYSA-N | $6.5\times10^{-7}$ $1.5\times10^{-6}$ $8.1\times10^{-7}$ | | Yaws (2003) Gharagheizi et al. (2012) Gharagheizi et al. (2010) | X Q Q | 237 246 |
| 2,2,4,4-tetramethyl-3-ethylpentane $C_{11}H_{24}$ [3178-30-1] FNWCWFQEEKFGHB-UHFFFAOYSA-N | $6.9\times10^{-7}$ $1.3\times10^{-6}$ $8.1\times10^{-7}$ | | Yaws (2003) Gharagheizi et al. (2012) Gharagheizi et al. (2010) | X Q Q | 237 246 |
| 2,2,4-trimethyl-3-isopropylpentane $C_{11}H_{24}$ [61868-90-4] SZPCILWZDHUKLW-UHFFFAOYSA-N | $8.7\times10^{-7}$ $9.4\times10^{-7}$ $9.0\times10^{-7}$ | | Yaws (2003) Gharagheizi et al. (2012) Gharagheizi et al. (2010) | X Q Q | 237 246 |
| 2,3,4-trimethyl-3-isopropylpentane $C_{11}H_{24}$ [61868-91-5] FHBFVPOXLRZZPP-UHFFFAOYSA-N | $7.1\times10^{-7}$ $1.5\times10^{-6}$ $9.0\times10^{-7}$ | | Yaws (2003) Gharagheizi et al. (2012) Gharagheizi et al. (2010) | X Q Q | 237 246 |





Table A2.1: Alkanes (. . . continued)

| Substance<br>Formula<br>(Trivial Name)<br>[CAS Registry Number]<br>InChIKey | $H_s^{cp}$<br>(at $T^\ominus$)<br>$\left[\dfrac{\text{mol}}{\text{m}^3\,\text{Pa}}\right]$ | $\dfrac{\text{d}\ln H_s^{cp}}{\text{d}(1/T)}$<br><br>[K] | Reference | Type | Note |
|---|---|---|---|---|---|
| 2,2-dimethyl-3,3-diethylpentane<br>$C_{11}H_{24}$<br>[60302-28-5]<br>KCQSYGZVYTYSLG-UHFFFAOYSA-N | $6.8\times10^{-7}$<br>$1.6\times10^{-6}$<br>$8.1\times10^{-7}$ | | Yaws (2003)<br>Gharagheizi et al. (2012)<br>Gharagheizi et al. (2010) | X<br>Q<br>Q | 237<br><br>246 |
| 2,4-dimethyl-3,3-diethylpentane<br>$C_{11}H_{24}$<br>[61868-92-6]<br>VVQOQVXGUMJNET-UHFFFAOYSA-N | $7.5\times10^{-7}$<br>$1.6\times10^{-6}$<br>$8.7\times10^{-7}$ | | Yaws (2003)<br>Gharagheizi et al. (2012)<br>Gharagheizi et al. (2010) | X<br>Q<br>Q | 237<br><br>246 |
| dodecane<br>$C_{12}H_{26}$<br>[112-40-3]<br>SNRUBQQJIBEYMU-UHFFFAOYSA-N | $1.2\times10^{-6}$<br>$1.3\times10^{-6}$<br>$1.2\times10^{-6}$<br>$1.2\times10^{-6}$<br>$1.2\times10^{-6}$<br>$1.3\times10^{-6}$<br>$1.2\times10^{-6}$<br>$1.2\times10^{-6}$<br>$1.2\times10^{-6}$<br>$3.0\times10^{-6}$<br>$4.4\times10^{-4}$<br>$1.7\times10^{-5}$<br>$8.5\times10^{-7}$<br>$5.3\times10^{-6}$<br>$5.1\times10^{-6}$<br>$1.2\times10^{-6}$<br>$1.6\times10^{-6}$<br>$9.9\times10^{-7}$<br>$1.6\times10^{-6}$<br>$1.1\times10^{-6}$<br>$3.1\times10^{-6}$<br>$1.2\times10^{-6}$<br>$1.4\times10^{-6}$ | | Plyasunov and Shock (2000)<br>Mackay and Shiu (1981)<br>Duchowicz et al. (2020)<br>HSDB (2015)<br>Mackay et al. (2006a)<br>Eastcott et al. (1988)<br>Abraham (1984)<br>Yaws (2003)<br>Yaws (2003)<br>Dupeux et al. (2022)<br>Duchowicz et al. (2020)<br>Wang et al. (2017)<br>Wang et al. (2017)<br>Wang et al. (2017)<br>Gharagheizi et al. (2012)<br>Raventos-Duran et al. (2010)<br>Raventos-Duran et al. (2010)<br>Raventos-Duran et al. (2010)<br>Gharagheizi et al. (2010)<br>Hilal et al. (2008)<br>Modarresi et al. (2007)<br>Yaws (1999)<br>Yaws and Yang (1992) | L<br>L<br>V<br>V<br>V<br>V<br>V<br>X<br>X<br>Q<br>Q<br>Q<br>Q<br>Q<br>Q<br>Q<br>Q<br>Q<br>Q<br>Q<br>Q<br>?<br>? | <br><br>186<br><br><br><br><br>258<br>237<br>259<br><br>80, 238<br>80, 239<br>80, 240<br><br>242, 243<br>244<br>245<br>246<br><br>67<br>21<br>21 |
| 2-methylundecane<br>$C_{12}H_{26}$<br>[31807-55-3]<br>GTJOHISYCKPIMT-UHFFFAOYSA-N | $1.1\times10^{-6}$<br>$4.2\times10^{-6}$<br>$1.3\times10^{-6}$ | | Yaws (2003)<br>Gharagheizi et al. (2012)<br>Gharagheizi et al. (2010) | X<br>Q<br>Q | 237<br><br>246 |
| 3-methylundecane<br>$C_{12}H_{26}$<br>[1002-43-3]<br>HTZWVZNRDDOFEI-UHFFFAOYSA-N | $1.1\times10^{-6}$<br>$4.2\times10^{-6}$<br>$1.3\times10^{-6}$ | | Yaws (2003)<br>Gharagheizi et al. (2012)<br>Gharagheizi et al. (2010) | X<br>Q<br>Q | 237<br><br>246 |
| 4-methylundecane<br>$C_{12}H_{26}$<br>[2980-69-0]<br>KNMXZGDUJVOTOC-UHFFFAOYSA-N | $1.1\times10^{-6}$<br>$2.7\times10^{-6}$<br>$1.3\times10^{-6}$ | | Yaws (2003)<br>Gharagheizi et al. (2012)<br>Gharagheizi et al. (2010) | X<br>Q<br>Q | 237<br><br>246 |



Table A2.1: Alkanes (... continued)

| Substance<br>Formula<br>(Trivial Name)<br>[CAS Registry Number]<br>InChIKey | $H_s^{cp}$<br>(at $T^{\ominus}$)<br>$\left[ \dfrac{\mathrm{mol}}{\mathrm{m^3\,Pa}} \right]$ | $\dfrac{\mathrm{d}\ln H_s^{cp}}{\mathrm{d}(1/T)}$<br><br>[K] | Reference | Type | Note |
|---|---|---|---|---|---|
| 5-methylundecane<br>$C_{12}H_{26}$<br>[1632-70-8]<br>QULNVKABFWNUCW-UHFFFAOYSA-N | $1.1\times10^{-6}$<br>$2.5\times10^{-6}$<br>$1.3\times10^{-6}$ | | Yaws (2003)<br>Gharagheizi et al. (2012)<br>Gharagheizi et al. (2010) | X<br>Q<br>Q | 237<br><br>246 |
| 6-methylundecane<br>$C_{12}H_{26}$<br>[17302-33-9]<br>VPYZCUCKYWHJGX-UHFFFAOYSA-N | $1.1\times10^{-6}$<br>$2.5\times10^{-6}$<br>$1.3\times10^{-6}$ | | Yaws (2003)<br>Gharagheizi et al. (2012)<br>Gharagheizi et al. (2010) | X<br>Q<br>Q | 237<br><br>246 |
| 2,2-dimethyldecane<br>$C_{12}H_{26}$<br>[17302-37-3]<br>WBWYXWILSHQILH-UHFFFAOYSA-N | $9.1\times10^{-7}$<br>$3.3\times10^{-6}$<br>$1.0\times10^{-6}$ | | Yaws (2003)<br>Gharagheizi et al. (2012)<br>Gharagheizi et al. (2010) | X<br>Q<br>Q | 237<br><br>246 |
| 2,3-dimethyldecane<br>$C_{12}H_{26}$<br>[17312-44-6]<br>ZCTGYLNFWOQVHV-UHFFFAOYSA-N | $1.0\times10^{-6}$<br>$2.5\times10^{-6}$<br>$1.0\times10^{-6}$ | | Yaws (2003)<br>Gharagheizi et al. (2012)<br>Gharagheizi et al. (2010) | X<br>Q<br>Q | 237<br><br>246 |
| 2,4-dimethyldecane<br>$C_{12}H_{26}$<br>[2801-84-5]<br>OJAFXEXESSNPMH-UHFFFAOYSA-N | $1.0\times10^{-6}$<br>$2.1\times10^{-6}$<br>$1.0\times10^{-6}$ | | Yaws (2003)<br>Gharagheizi et al. (2012)<br>Gharagheizi et al. (2010) | X<br>Q<br>Q | 237<br><br>246 |
| 2,5-dimethyldecane<br>$C_{12}H_{26}$<br>[17312-50-4]<br>DQHKBYZSYRJBMD-UHFFFAOYSA-N | $1.0\times10^{-6}$<br>$2.0\times10^{-6}$<br>$1.0\times10^{-6}$ | | Yaws (2003)<br>Gharagheizi et al. (2012)<br>Gharagheizi et al. (2010) | X<br>Q<br>Q | 237<br><br>246 |
| 2,6-dimethyldecane<br>$C_{12}H_{26}$<br>[13150-81-7]<br>DHJGXZWEQBKLNV-UHFFFAOYSA-N | $1.0\times10^{-6}$<br>$2.0\times10^{-6}$<br>$1.0\times10^{-6}$ | | Yaws (2003)<br>Gharagheizi et al. (2012)<br>Gharagheizi et al. (2010) | X<br>Q<br>Q | 237<br><br>246 |
| 2,7-dimethyldecane<br>$C_{12}H_{26}$<br>[17312-51-5]<br>RVQIXUWWPOTVNP-UHFFFAOYSA-N | $1.0\times10^{-6}$<br>$2.2\times10^{-6}$<br>$1.0\times10^{-6}$ | | Yaws (2003)<br>Gharagheizi et al. (2012)<br>Gharagheizi et al. (2010) | X<br>Q<br>Q | 237<br><br>246 |
| 2,8-dimethyldecane<br>$C_{12}H_{26}$<br>[17312-52-6]<br>KSRGGHUVCVWVDW-UHFFFAOYSA-N | $1.0\times10^{-6}$<br>$2.3\times10^{-6}$<br>$1.0\times10^{-6}$ | | Yaws (2003)<br>Gharagheizi et al. (2012)<br>Gharagheizi et al. (2010) | X<br>Q<br>Q | 237<br><br>246 |
| 2,9-dimethyldecane<br>$C_{12}H_{26}$<br>[1002-17-1]<br>HWISDPDDDUZJAW-UHFFFAOYSA-N | $1.0\times10^{-6}$<br>$5.5\times10^{-6}$<br>$1.0\times10^{-6}$ | | Yaws (2003)<br>Gharagheizi et al. (2012)<br>Gharagheizi et al. (2010) | X<br>Q<br>Q | 237<br><br>246 |





Table A2.1: Alkanes (... continued)

| Substance<br>Formula<br>(Trivial Name)<br>[CAS Registry Number]<br>InChIKey | $H_s^{cp}$<br>(at $T^\ominus$)<br>$\left[\dfrac{\mathrm{mol}}{\mathrm{m^3\,Pa}}\right]$ | $\dfrac{\mathrm{d}\ln H_s^{cp}}{\mathrm{d}(1/T)}$<br><br>[K] | Reference | Type | Note |
|---|---|---|---|---|---|
| 3,3-dimethyldecane<br>$C_{12}H_{26}$<br>[17302-38-4]<br>URERYYDSQOIHQK-UHFFFAOYSA-N | $9.1\times10^{-7}$<br>$1.5\times10^{-6}$<br>$1.0\times10^{-6}$ | | Yaws (2003)<br>Gharagheizi et al. (2012)<br>Gharagheizi et al. (2010) | X<br>Q<br>Q | 237<br><br>246 |
| 3,4-dimethyldecane<br>$C_{12}H_{26}$<br>[17312-45-7]<br>NRBMEEDORZDRIT-UHFFFAOYSA-N | $1.0\times10^{-6}$<br>$1.6\times10^{-6}$<br>$1.0\times10^{-6}$ | | Yaws (2003)<br>Gharagheizi et al. (2012)<br>Gharagheizi et al. (2010) | X<br>Q<br>Q | 237<br><br>246 |
| 3,5-dimethyldecane<br>$C_{12}H_{26}$<br>[17312-48-0]<br>XXSUEVGKGOUJMD-UHFFFAOYSA-N | $1.0\times10^{-6}$<br>$1.4\times10^{-6}$<br>$1.0\times10^{-6}$ | | Yaws (2003)<br>Gharagheizi et al. (2012)<br>Gharagheizi et al. (2010) | X<br>Q<br>Q | 237<br><br>246 |
| 3,6-dimethyldecane<br>$C_{12}H_{26}$<br>[17312-53-7]<br>NQWFSCYWTXQNGG-UHFFFAOYSA-N | $1.0\times10^{-6}$<br>$1.4\times10^{-6}$<br>$1.0\times10^{-6}$ | | Yaws (2003)<br>Gharagheizi et al. (2012)<br>Gharagheizi et al. (2010) | X<br>Q<br>Q | 237<br><br>246 |
| 3,7-dimethyldecane<br>$C_{12}H_{26}$<br>[17312-54-8]<br>VDAVEASVPZDNQB-UHFFFAOYSA-N | $1.0\times10^{-6}$<br>$1.5\times10^{-6}$<br>$1.0\times10^{-6}$ | | Yaws (2003)<br>Gharagheizi et al. (2012)<br>Gharagheizi et al. (2010) | X<br>Q<br>Q | 237<br><br>246 |
| 3,8-dimethyldecane<br>$C_{12}H_{26}$<br>[17312-55-9]<br>KMAHIPNGGSOJSM-UHFFFAOYSA-N | $1.0\times10^{-6}$<br>$1.6\times10^{-6}$<br>$1.0\times10^{-6}$ | | Yaws (2003)<br>Gharagheizi et al. (2012)<br>Gharagheizi et al. (2010) | X<br>Q<br>Q | 237<br><br>246 |
| 4,4-dimethyldecane<br>$C_{12}H_{26}$<br>[17312-39-9]<br>WZCACTKWHXCWFZ-UHFFFAOYSA-N | $9.2\times10^{-7}$<br>$1.3\times10^{-6}$<br>$1.0\times10^{-6}$ | | Yaws (2003)<br>Gharagheizi et al. (2012)<br>Gharagheizi et al. (2010) | X<br>Q<br>Q | 237<br><br>246 |
| 4,5-dimethyldecane<br>$C_{12}H_{26}$<br>[17312-46-8]<br>QZFIIEYSHODCSV-UHFFFAOYSA-N | $1.0\times10^{-6}$<br>$1.5\times10^{-6}$<br>$1.0\times10^{-6}$ | | Yaws (2003)<br>Gharagheizi et al. (2012)<br>Gharagheizi et al. (2010) | X<br>Q<br>Q | 237<br><br>246 |
| 4,6-dimethyldecane<br>$C_{12}H_{26}$<br>[17312-49-1]<br>TVGNRLXIJXKVGD-UHFFFAOYSA-N | $1.0\times10^{-6}$<br>$1.3\times10^{-6}$<br>$1.0\times10^{-6}$ | | Yaws (2003)<br>Gharagheizi et al. (2012)<br>Gharagheizi et al. (2010) | X<br>Q<br>Q | 237<br><br>246 |
| 4,7-dimethyldecane<br>$C_{12}H_{26}$<br>[17312-56-0]<br>GCKWUFQALHAZDH-UHFFFAOYSA-N | $1.0\times10^{-6}$<br>$1.3\times10^{-6}$<br>$1.0\times10^{-6}$ | | Yaws (2003)<br>Gharagheizi et al. (2012)<br>Gharagheizi et al. (2010) | X<br>Q<br>Q | 237<br><br>246 |



Table A2.1: Alkanes (...continued)

| Substance Formula (Trivial Name) [CAS Registry Number] InChIKey | $H_s^{cp}$ (at $T^{\ominus}$) $\left[\dfrac{\text{mol}}{\text{m}^3\,\text{Pa}}\right]$ | $\dfrac{\text{d}\ln H_s^{cp}}{\text{d}(1/T)}$ [K] | Reference | Type | Note |
|---|---|---|---|---|---|
| 5,5-dimethyldecane $C_{12}H_{26}$ [17453-92-8] RNXSOUOIPAWOAC-UHFFFAOYSA-N | $9.3\times10^{-7}$ $1.3\times10^{-6}$ $1.0\times10^{-6}$ | | Yaws (2003) Gharagheizi et al. (2012) Gharagheizi et al. (2010) | X Q Q | 237 246 |
| 5,6-dimethyldecane $C_{12}H_{26}$ [1636-43-7] NCJIZIYQFWXMFZ-UHFFFAOYSA-N | $1.0\times10^{-6}$ $1.4\times10^{-6}$ $1.0\times10^{-6}$ | | Yaws (2003) Gharagheizi et al. (2012) Gharagheizi et al. (2010) | X Q Q | 237 246 |
| 3-ethyldecane $C_{12}H_{26}$ [17085-96-0] ZBDDVSBBCGZQDV-UHFFFAOYSA-N | $1.1\times10^{-6}$ $2.7\times10^{-6}$ $1.3\times10^{-6}$ | | Yaws (2003) Gharagheizi et al. (2012) Gharagheizi et al. (2010) | X Q Q | 237 246 |
| 4-ethyldecane $C_{12}H_{26}$ [1636-44-8] IGTKVLJTIZALGL-UHFFFAOYSA-N | $1.1\times10^{-6}$ $1.5\times10^{-6}$ $1.3\times10^{-6}$ | | Yaws (2003) Gharagheizi et al. (2012) Gharagheizi et al. (2010) | X Q Q | 237 246 |
| 5-ethyldecane $C_{12}H_{26}$ [17302-36-2] BCQLAHKMMOGIIS-UHFFFAOYSA-N | $1.1\times10^{-6}$ $1.4\times10^{-6}$ $1.3\times10^{-6}$ | | Yaws (2003) Gharagheizi et al. (2012) Gharagheizi et al. (2010) | X Q Q | 237 246 |
| 2,2,3-trimethylnonane $C_{12}H_{26}$ [55499-04-2] XXNUJUNKYOZLAJ-UHFFFAOYSA-N | $8.3\times10^{-7}$ $9.2\times10^{-7}$ $8.7\times10^{-7}$ | | Yaws (2003) Gharagheizi et al. (2012) Gharagheizi et al. (2010) | X Q Q | 237 246 |
| 2,2,4-trimethylnonane $C_{12}H_{26}$ [62184-50-3] YYYAICGYFLHBIN-UHFFFAOYSA-N | $8.8\times10^{-7}$ $7.1\times10^{-7}$ $8.7\times10^{-7}$ | | Yaws (2003) Gharagheizi et al. (2012) Gharagheizi et al. (2010) | X Q Q | 237 246 |
| 2,2,5-trimethylnonane $C_{12}H_{26}$ [62184-51-4] ZIXRFABUOOUIMA-UHFFFAOYSA-N | $8.8\times10^{-7}$ $7.1\times10^{-7}$ $8.7\times10^{-7}$ | | Yaws (2003) Gharagheizi et al. (2012) Gharagheizi et al. (2010) | X Q Q | 237 246 |
| 2,2,6-trimethylnonane $C_{12}H_{26}$ [62184-52-5] BUTMZMIZSWINDK-UHFFFAOYSA-N | $8.8\times10^{-7}$ $6.9\times10^{-7}$ $8.7\times10^{-7}$ | | Yaws (2003) Gharagheizi et al. (2012) Gharagheizi et al. (2010) | X Q Q | 237 246 |
| 2,2,7-trimethylnonane $C_{12}H_{26}$ [62184-53-6] PXOURNMYVAOUML-UHFFFAOYSA-N | $8.5\times10^{-7}$ $7.9\times10^{-7}$ $8.7\times10^{-7}$ | | Yaws (2003) Gharagheizi et al. (2012) Gharagheizi et al. (2010) | X Q Q | 237 246 |



Table A2.1: Alkanes (...continued)

| Substance Formula (Trivial Name) [CAS Registry Number] InChIKey | $H_s^{cp}$ (at $T^\ominus$) $\left[\dfrac{\mathrm{mol}}{\mathrm{m^3\,Pa}}\right]$ | $\dfrac{\mathrm{d}\ln H_s^{cp}}{\mathrm{d}(1/T)}$ [K] | Reference | Type | Note |
|---|---|---|---|---|---|
| 2,2,8-trimethylnonane | $8.5\times10^{-7}$ | | Yaws (2003) | X | 237 |
| $C_{12}H_{26}$ | $7.9\times10^{-7}$ | | Gharagheizi et al. (2012) | Q | |
| [62184-54-7] | $8.7\times10^{-7}$ | | Gharagheizi et al. (2010) | Q | 246 |
| WXUAVABOZJOELJ-UHFFFAOYSA-N | | | | | |
| 2,3,3-trimethylnonane | $8.3\times10^{-7}$ | | Yaws (2003) | X | 237 |
| $C_{12}H_{26}$ | $9.7\times10^{-7}$ | | Gharagheizi et al. (2012) | Q | |
| [62184-55-8] | $8.7\times10^{-7}$ | | Gharagheizi et al. (2010) | Q | 246 |
| RSUFJUIYLXRYFG-UHFFFAOYSA-N | | | | | |
| 2,3,4-trimethylnonane | $9.2\times10^{-7}$ | | Yaws (2003) | X | 237 |
| $C_{12}H_{26}$ | $9.1\times10^{-7}$ | | Gharagheizi et al. (2012) | Q | |
| [62184-56-9] | $9.2\times10^{-7}$ | | Gharagheizi et al. (2010) | Q | 246 |
| RRZRZLRUYPZYEH-UHFFFAOYSA-N | | | | | |
| 2,3,5-trimethylnonane | $9.4\times10^{-7}$ | | Yaws (2003) | X | 237 |
| $C_{12}H_{26}$ | $8.1\times10^{-7}$ | | Gharagheizi et al. (2012) | Q | |
| [62184-57-0] | $9.2\times10^{-7}$ | | Gharagheizi et al. (2010) | Q | 246 |
| IESSJVKAXJWJFD-UHFFFAOYSA-N | | | | | |
| 2,3,6-trimethylnonane | $9.3\times10^{-7}$ | | Yaws (2003) | X | 237 |
| $C_{12}H_{26}$ | $8.6\times10^{-7}$ | | Gharagheizi et al. (2012) | Q | |
| [62184-58-1] | $9.2\times10^{-7}$ | | Gharagheizi et al. (2010) | Q | 246 |
| AUXUELQPRQTQBE-UHFFFAOYSA-N | | | | | |
| 2,3,7-trimethylnonane | $9.2\times10^{-7}$ | | Yaws (2003) | X | 237 |
| $C_{12}H_{26}$ | $9.4\times10^{-7}$ | | Gharagheizi et al. (2012) | Q | |
| [62184-59-2] | $9.2\times10^{-7}$ | | Gharagheizi et al. (2010) | Q | 246 |
| GGUGBCMHEULHJS-UHFFFAOYSA-N | | | | | |
| 2,3,8-trimethylnonane | $9.2\times10^{-7}$ | | Yaws (2003) | X | 237 |
| $C_{12}H_{26}$ | $9.1\times10^{-7}$ | | Gharagheizi et al. (2012) | Q | |
| [62184-60-5] | $9.2\times10^{-7}$ | | Gharagheizi et al. (2010) | Q | 246 |
| GQHCJRDJABETBJ-UHFFFAOYSA-N | | | | | |
| 2,4,4-trimethylnonane | $8.8\times10^{-7}$ | | Yaws (2003) | X | 237 |
| $C_{12}H_{26}$ | $6.9\times10^{-7}$ | | Gharagheizi et al. (2012) | Q | |
| [62184-61-6] | $8.7\times10^{-7}$ | | Gharagheizi et al. (2010) | Q | 246 |
| RRLKOZCPQXMIIJ-UHFFFAOYSA-N | | | | | |
| 2,4,5-trimethylnonane | $9.4\times10^{-7}$ | | Yaws (2003) | X | 237 |
| $C_{12}H_{26}$ | $7.9\times10^{-7}$ | | Gharagheizi et al. (2012) | Q | |
| [62184-62-7] | $9.2\times10^{-7}$ | | Gharagheizi et al. (2010) | Q | 246 |
| NZAPYIGNIYRJNJ-UHFFFAOYSA-N | | | | | |
| 2,4,6-trimethylnonane | $9.6\times10^{-7}$ | | Yaws (2003) | X | 237 |
| $C_{12}H_{26}$ | $7.2\times10^{-7}$ | | Gharagheizi et al. (2012) | Q | |
| [62184-10-5] | $9.2\times10^{-7}$ | | Gharagheizi et al. (2010) | Q | 246 |
| QCQMRLDBNMVKOZ-UHFFFAOYSA-N | | | | | |





Table A2.1: Alkanes (...continued)

| Substance<br>Formula<br>(Trivial Name)<br>[CAS Registry Number]<br>InChIKey | $H_s^{cp}$<br>(at $T^{\ominus}$)<br>$\left[\dfrac{\text{mol}}{\text{m}^3\,\text{Pa}}\right]$ | $\dfrac{\text{d}\ln H_s^{cp}}{\text{d}(1/T)}$<br><br>[K] | Reference | Type | Note |
|---|---|---|---|---|---|
| 2,4,7-trimethylnonane<br>$C_{12}H_{26}$<br>[62184-11-6]<br>OHBDIVOYUFLOTO-UHFFFAOYSA-N | $9.5\times10^{-7}$<br>$7.7\times10^{-7}$<br>$9.2\times10^{-7}$ | | Yaws (2003)<br>Gharagheizi et al. (2012)<br>Gharagheizi et al. (2010) | X<br>Q<br>Q | 237<br><br>246 |
| 2,4,8-trimethylnonane<br>$C_{12}H_{26}$<br>[49542-74-7]<br>PZXMOJBYRFSNJH-UHFFFAOYSA-N | $9.6\times10^{-7}$<br>$7.0\times10^{-7}$<br>$9.2\times10^{-7}$ | | Yaws (2003)<br>Gharagheizi et al. (2012)<br>Gharagheizi et al. (2010) | X<br>Q<br>Q | 237<br><br>246 |
| 2,5,5-trimethylnonane<br>$C_{12}H_{26}$<br>[62184-12-7]<br>ZRRKWMYBHUMSTG-UHFFFAOYSA-N | $8.8\times10^{-7}$<br>$6.9\times10^{-7}$<br>$8.7\times10^{-7}$ | | Yaws (2003)<br>Gharagheizi et al. (2012)<br>Gharagheizi et al. (2010) | X<br>Q<br>Q | 237<br><br>246 |
| 2,5,6-trimethylnonane<br>$C_{12}H_{26}$<br>[62184-13-8]<br>KXHDWXIQHWSOHI-UHFFFAOYSA-N | $9.4\times10^{-7}$<br>$8.1\times10^{-7}$<br>$9.2\times10^{-7}$ | | Yaws (2003)<br>Gharagheizi et al. (2012)<br>Gharagheizi et al. (2010) | X<br>Q<br>Q | 237<br><br>246 |
| 2,5,7-trimethylnonane<br>$C_{12}H_{26}$<br>[62184-14-9]<br>KMBPYTMBXFKXON-UHFFFAOYSA-N | $9.5\times10^{-7}$<br>$7.7\times10^{-7}$<br>$9.2\times10^{-7}$ | | Yaws (2003)<br>Gharagheizi et al. (2012)<br>Gharagheizi et al. (2010) | X<br>Q<br>Q | 237<br><br>246 |
| 2,5,8-trimethylnonane<br>$C_{12}H_{26}$<br>[49557-09-7]<br>YYZRNFMGAQMLJJ-UHFFFAOYSA-N | $9.6\times10^{-7}$<br>$7.2\times10^{-7}$<br>$9.2\times10^{-7}$ | | Yaws (2003)<br>Gharagheizi et al. (2012)<br>Gharagheizi et al. (2010) | X<br>Q<br>Q | 237<br><br>246 |
| 2,6,6-trimethylnonane<br>$C_{12}H_{26}$<br>[62184-15-0]<br>MTPVPWZHWOHMQT-UHFFFAOYSA-N | $8.8\times10^{-7}$<br>$6.9\times10^{-7}$<br>$8.7\times10^{-7}$ | | Yaws (2003)<br>Gharagheizi et al. (2012)<br>Gharagheizi et al. (2010) | X<br>Q<br>Q | 237<br><br>246 |
| 2,6,7-trimethylnonane<br>$C_{12}H_{26}$<br>[62184-16-1]<br>PGLYQCHTCUBASV-UHFFFAOYSA-N | $9.3\times10^{-7}$<br>$8.9\times10^{-7}$<br>$9.2\times10^{-7}$ | | Yaws (2003)<br>Gharagheizi et al. (2012)<br>Gharagheizi et al. (2010) | X<br>Q<br>Q | 237<br><br>246 |
| 2,7,7-trimethylnonane<br>$C_{12}H_{26}$<br>[62184-17-2]<br>RCTMUTWYOMEASD-UHFFFAOYSA-N | $8.5\times10^{-7}$<br>$8.4\times10^{-7}$<br>$8.7\times10^{-7}$ | | Yaws (2003)<br>Gharagheizi et al. (2012)<br>Gharagheizi et al. (2010) | X<br>Q<br>Q | 237<br><br>246 |
| 3,3,4-trimethylnonane<br>$C_{12}H_{26}$<br>[62184-18-3]<br>RNCRKMFELJSFRA-UHFFFAOYSA-N | $8.3\times10^{-7}$<br>$9.7\times10^{-7}$<br>$8.7\times10^{-7}$ | | Yaws (2003)<br>Gharagheizi et al. (2012)<br>Gharagheizi et al. (2010) | X<br>Q<br>Q | 237<br><br>246 |



Table A2.1: Alkanes (. . . continued)

| Substance Formula (Trivial Name) [CAS Registry Number] InChIKey | $H_s^{cp}$ (at $T^{\ominus}$) $\left[\dfrac{\mathrm{mol}}{\mathrm{m}^3\,\mathrm{Pa}}\right]$ | $\dfrac{\mathrm{d}\ln H_s^{cp}}{\mathrm{d}(1/T)}$ [K] | Reference | Type | Note |
|---|---|---|---|---|---|
| 3,3,5-trimethylnonane $C_{12}H_{26}$ [62184-19-4] KRWQPUYFIGHZNF-UHFFFAOYSA-N | $8.8\times10^{-7}$ $6.9\times10^{-7}$ $8.7\times10^{-7}$ | | Yaws (2003) Gharagheizi et al. (2012) Gharagheizi et al. (2010) | X Q Q | 237 246 |
| 3,3,6-trimethylnonane $C_{12}H_{26}$ [62184-20-7] GPSMOBMYMZBKGT-UHFFFAOYSA-N | $8.4\times10^{-7}$ $8.6\times10^{-7}$ $8.7\times10^{-7}$ | | Yaws (2003) Gharagheizi et al. (2012) Gharagheizi et al. (2010) | X Q Q | 237 246 |
| 3,3,7-trimethylnonane $C_{12}H_{26}$ [62184-21-8] NZYGFHSHEQXPFT-UHFFFAOYSA-N | $8.5\times10^{-7}$ $7.9\times10^{-7}$ $8.7\times10^{-7}$ | | Yaws (2003) Gharagheizi et al. (2012) Gharagheizi et al. (2010) | X Q Q | 237 246 |
| 3,4,4-trimethylnonane $C_{12}H_{26}$ [62184-22-9] UYNWXBDPBKCTRS-UHFFFAOYSA-N | $8.3\times10^{-7}$ $9.2\times10^{-7}$ $8.7\times10^{-7}$ | | Yaws (2003) Gharagheizi et al. (2012) Gharagheizi et al. (2010) | X Q Q | 237 246 |
| 3,4,5-trimethylnonane $C_{12}H_{26}$ [62184-23-0] PFVDUOFPBLPHEQ-UHFFFAOYSA-N | $9.2\times10^{-7}$ $9.1\times10^{-7}$ $9.2\times10^{-7}$ | | Yaws (2003) Gharagheizi et al. (2012) Gharagheizi et al. (2010) | X Q Q | 237 246 |
| 3,4,6-trimethylnonane $C_{12}H_{26}$ [62184-24-1] VSRUKIVBXBGRDU-UHFFFAOYSA-N | $9.3\times10^{-7}$ $8.4\times10^{-7}$ $9.2\times10^{-7}$ | | Yaws (2003) Gharagheizi et al. (2012) Gharagheizi et al. (2010) | X Q Q | 237 246 |
| 3,4,7-trimethylnonane $C_{12}H_{26}$ [27802-85-3] HQIBDLHPQGDJEW-UHFFFAOYSA-N | $9.2\times10^{-7}$ $9.1\times10^{-7}$ $9.2\times10^{-7}$ | | Yaws (2003) Gharagheizi et al. (2012) Gharagheizi et al. (2010) | X Q Q | 237 246 |
| 3,5,5-trimethylnonane $C_{12}H_{26}$ [62184-25-2] MUVSDRJGHVLJQX-UHFFFAOYSA-N | $8.8\times10^{-7}$ $7.1\times10^{-7}$ $8.7\times10^{-7}$ | | Yaws (2003) Gharagheizi et al. (2012) Gharagheizi et al. (2010) | X Q Q | 237 246 |
| 3,5,6-trimethylnonane $C_{12}H_{26}$ [62184-26-3] HMCPPWBQUPZETF-UHFFFAOYSA-N | $9.3\times10^{-7}$ $8.4\times10^{-7}$ $9.2\times10^{-7}$ | | Yaws (2003) Gharagheizi et al. (2012) Gharagheizi et al. (2010) | X Q Q | 237 246 |
| 3,5,7-trimethylnonane $C_{12}H_{26}$ [62184-27-4] LLVWEFHXQBAEJX-UHFFFAOYSA-N | $9.4\times10^{-7}$ $7.9\times10^{-7}$ $9.2\times10^{-7}$ | | Yaws (2003) Gharagheizi et al. (2012) Gharagheizi et al. (2010) | X Q Q | 237 246 |



Table A2.1: Alkanes (. . . continued)

| Substance<br>Formula<br>(Trivial Name)<br>[CAS Registry Number]<br>InChIKey | $H_s^{cp}$<br>(at $T^\ominus$)<br>$\left[\dfrac{\text{mol}}{\text{m}^3\,\text{Pa}}\right]$ | $\dfrac{\mathrm{d}\ln H_s^{cp}}{\mathrm{d}(1/T)}$<br><br>[K] | Reference | Type | Note |
|---|---|---|---|---|---|
| 3,6,6-trimethylnonane<br>$C_{12}H_{26}$<br>[62184-28-5]<br>CBDYLMAJISNHBH-UHFFFAOYSA-N | $8.8\times10^{-7}$<br>$6.9\times10^{-7}$<br>$8.7\times10^{-7}$ | | Yaws (2003)<br>Gharagheizi et al. (2012)<br>Gharagheizi et al. (2010) | X<br>Q<br>Q | 237<br><br>246 |
| 4,4,5-trimethylnonane<br>$C_{12}H_{26}$<br>[62184-29-6]<br>QLJZSYCAASGFGU-UHFFFAOYSA-N | $8.4\times10^{-7}$<br>$8.6\times10^{-7}$<br>$8.7\times10^{-7}$ | | Yaws (2003)<br>Gharagheizi et al. (2012)<br>Gharagheizi et al. (2010) | X<br>Q<br>Q | 237<br><br>246 |
| 4,4,6-trimethylnonane<br>$C_{12}H_{26}$<br>[62184-30-9]<br>KBTTWWLCACXOSV-UHFFFAOYSA-N | $8.8\times10^{-7}$<br>$7.1\times10^{-7}$<br>$8.7\times10^{-7}$ | | Yaws (2003)<br>Gharagheizi et al. (2012)<br>Gharagheizi et al. (2010) | X<br>Q<br>Q | 237<br><br>246 |
| 4,5,5-trimethylnonane<br>$C_{12}H_{26}$<br>[62184-31-0]<br>QVDLUCCOJXIINS-UHFFFAOYSA-N | $8.4\times10^{-7}$<br>$8.6\times10^{-7}$<br>$8.7\times10^{-7}$ | | Yaws (2003)<br>Gharagheizi et al. (2012)<br>Gharagheizi et al. (2010) | X<br>Q<br>Q | 237<br><br>246 |
| 4,5,6-trimethylnonane<br>$C_{12}H_{26}$<br>[62211-85-2]<br>BBEKERHGCOZOIX-UHFFFAOYSA-N | $9.3\times10^{-7}$<br>$8.6\times10^{-7}$<br>$9.2\times10^{-7}$ | | Yaws (2003)<br>Gharagheizi et al. (2012)<br>Gharagheizi et al. (2010) | X<br>Q<br>Q | 237<br><br>246 |
| 2-methyl-3-ethylnonane<br>$C_{12}H_{26}$<br>[62184-73-0]<br>NVCDNLWOUSWNRQ-UHFFFAOYSA-N | $1.0\times10^{-6}$<br>$9.3\times10^{-7}$<br>$1.0\times10^{-6}$ | | Yaws (2003)<br>Gharagheizi et al. (2012)<br>Gharagheizi et al. (2010) | X<br>Q<br>Q | 237<br><br>246 |
| 2-methyl-4-ethylnonane<br>$C_{12}H_{26}$<br>[62184-37-6]<br>GRYPXYDXCQOVAY-UHFFFAOYSA-N | $1.0\times10^{-6}$<br>$8.5\times10^{-7}$<br>$1.0\times10^{-6}$ | | Yaws (2003)<br>Gharagheizi et al. (2012)<br>Gharagheizi et al. (2010) | X<br>Q<br>Q | 237<br><br>246 |
| 2-methyl-5-ethylnonane<br>$C_{12}H_{26}$<br>[62184-38-7]<br>WZWWJMDOQQIPJZ-UHFFFAOYSA-N | $1.0\times10^{-6}$<br>$8.1\times10^{-7}$<br>$1.0\times10^{-6}$ | | Yaws (2003)<br>Gharagheizi et al. (2012)<br>Gharagheizi et al. (2010) | X<br>Q<br>Q | 237<br><br>246 |
| 2-methyl-6-ethylnonane<br>$C_{12}H_{26}$<br>[62184-39-8]<br>GBMUYXDAAUUNLY-UHFFFAOYSA-N | $1.0\times10^{-6}$<br>$8.5\times10^{-7}$<br>$1.0\times10^{-6}$ | | Yaws (2003)<br>Gharagheizi et al. (2012)<br>Gharagheizi et al. (2010) | X<br>Q<br>Q | 237<br><br>246 |
| 2-methyl-7-ethylnonane<br>$C_{12}H_{26}$<br>[62184-40-1]<br>PHPGTUZXJVCCKP-UHFFFAOYSA-N | $1.0\times10^{-6}$<br>$9.6\times10^{-7}$<br>$1.0\times10^{-6}$ | | Yaws (2003)<br>Gharagheizi et al. (2012)<br>Gharagheizi et al. (2010) | X<br>Q<br>Q | 237<br><br>246 |





Table A2.1: Alkanes (...continued)

| Substance Formula (Trivial Name) [CAS Registry Number] InChIKey | $H_s^{cp}$ (at $T^\ominus$) $\left[\dfrac{\text{mol}}{\text{m}^3\,\text{Pa}}\right]$ | $\dfrac{\text{d}\ln H_s^{cp}}{\text{d}(1/T)}$ [K] | Reference | Type | Note |
|---|---|---|---|---|---|
| 3-methyl-3-ethylnonane | $9.1\times10^{-7}$ | | Yaws (2003) | X | 237 |
| $C_{12}H_{26}$ | $1.0\times10^{-6}$ | | Gharagheizi et al. (2012) | Q | |
| [17302-39-5] | $1.0\times10^{-6}$ | | Gharagheizi et al. (2010) | Q | 246 |
| MJZXPSBAAXOUJF-UHFFFAOYSA-N | | | | | |
| 3-methyl-4-ethylnonane | $1.0\times10^{-6}$ | | Yaws (2003) | X | 237 |
| $C_{12}H_{26}$ | $9.3\times10^{-7}$ | | Gharagheizi et al. (2012) | Q | |
| [62184-41-2] | $1.0\times10^{-6}$ | | Gharagheizi et al. (2010) | Q | 246 |
| ZELDGOVWPZHVJS-UHFFFAOYSA-N | | | | | |
| 3-methyl-5-ethylnonane | $1.0\times10^{-6}$ | | Yaws (2003) | X | 237 |
| $C_{12}H_{26}$ | $7.8\times10^{-7}$ | | Gharagheizi et al. (2012) | Q | |
| [62184-42-3] | $1.0\times10^{-6}$ | | Gharagheizi et al. (2010) | Q | 246 |
| GDUZVOIDJDZQBC-UHFFFAOYSA-N | | | | | |
| 3-methyl-6-ethylnonane | $1.0\times10^{-6}$ | | Yaws (2003) | X | 237 |
| $C_{12}H_{26}$ | $8.8\times10^{-7}$ | | Gharagheizi et al. (2012) | Q | |
| [62184-43-4] | $1.0\times10^{-6}$ | | Gharagheizi et al. (2010) | Q | 246 |
| WSMCXXNPFLZSMY-UHFFFAOYSA-N | | | | | |
| 3-methyl-7-ethylnonane | $1.0\times10^{-6}$ | | Yaws (2003) | X | 237 |
| $C_{12}H_{26}$ | $9.0\times10^{-7}$ | | Gharagheizi et al. (2012) | Q | |
| [62184-44-5] | $1.0\times10^{-6}$ | | Gharagheizi et al. (2010) | Q | 246 |
| FSGFZSLYTXQNCW-UHFFFAOYSA-N | | | | | |
| 4-methyl-3-ethylnonane | $1.0\times10^{-6}$ | | Yaws (2003) | X | 237 |
| $C_{12}H_{26}$ | $9.0\times10^{-7}$ | | Gharagheizi et al. (2012) | Q | |
| [62184-45-6] | $1.0\times10^{-6}$ | | Gharagheizi et al. (2010) | Q | 246 |
| AUMKRFNJSTUECA-UHFFFAOYSA-N | | | | | |
| 4-methyl-4-ethylnonane | $9.2\times10^{-7}$ | | Yaws (2003) | X | 237 |
| $C_{12}H_{26}$ | $8.8\times10^{-7}$ | | Gharagheizi et al. (2012) | Q | |
| [17312-40-2] | $1.0\times10^{-6}$ | | Gharagheizi et al. (2010) | Q | 246 |
| LXQCGRFCISXKCQ-UHFFFAOYSA-N | | | | | |
| 4-methyl-5-ethylnonane | $1.0\times10^{-6}$ | | Yaws (2003) | X | 237 |
| $C_{12}H_{26}$ | $8.5\times10^{-7}$ | | Gharagheizi et al. (2012) | Q | |
| [62184-46-7] | $1.0\times10^{-6}$ | | Gharagheizi et al. (2010) | Q | 246 |
| BDJIXOFXGLDYLN-UHFFFAOYSA-N | | | | | |
| 4-methyl-6-ethylnonane | $1.0\times10^{-6}$ | | Yaws (2003) | X | 237 |
| $C_{12}H_{26}$ | $8.1\times10^{-7}$ | | Gharagheizi et al. (2012) | Q | |
| [62184-47-8] | $1.0\times10^{-6}$ | | Gharagheizi et al. (2010) | Q | 246 |
| SULZKBWLMOBACQ-UHFFFAOYSA-N | | | | | |
| 4-methyl-7-ethylnonane | $1.0\times10^{-6}$ | | Yaws (2003) | X | 237 |
| $C_{12}H_{26}$ | $9.0\times10^{-7}$ | | Gharagheizi et al. (2012) | Q | |
| [62184-48-9] | $1.0\times10^{-6}$ | | Gharagheizi et al. (2010) | Q | 246 |
| HYIBWPIYSSTZLB-UHFFFAOYSA-N | | | | | |





Table A2.1: Alkanes (...continued)

| Substance<br>Formula<br>(Trivial Name)<br>[CAS Registry Number]<br>InChIKey | $H_s^{cp}$<br>(at $T^{\ominus}$)<br>$\left[\dfrac{\mathrm{mol}}{\mathrm{m^3\,Pa}}\right]$ | $\dfrac{\mathrm{d}\ln H_s^{cp}}{\mathrm{d}(1/T)}$<br><br>[K] | Reference | Type | Note |
|---|---|---|---|---|---|
| 5-methyl-3-ethylnonane<br>$C_{12}H_{26}$<br>[62184-49-0]<br>ZBALIHDUORTBQR-UHFFFAOYSA-N | $1.0\times10^{-6}$<br>$8.8\times10^{-7}$<br>$1.0\times10^{-6}$ | | Yaws (2003)<br>Gharagheizi et al. (2012)<br>Gharagheizi et al. (2010) | X<br>Q<br>Q | 237<br><br>246 |
| 5-methyl-4-ethylnonane<br>$C_{12}H_{26}$<br>[1632-71-9]<br>HKUUNAOHRKAJIY-UHFFFAOYSA-N | $1.0\times10^{-6}$<br>$8.8\times10^{-7}$<br>$1.0\times10^{-6}$ | | Yaws (2003)<br>Gharagheizi et al. (2012)<br>Gharagheizi et al. (2010) | X<br>Q<br>Q | 237<br><br>246 |
| 5-methyl-5-ethylnonane<br>$C_{12}H_{26}$<br>[14531-16-9]<br>DZSBQMNPPIKBNF-UHFFFAOYSA-N | $9.2\times10^{-7}$<br>$8.6\times10^{-7}$<br>$1.0\times10^{-6}$ | | Yaws (2003)<br>Gharagheizi et al. (2012)<br>Gharagheizi et al. (2010) | X<br>Q<br>Q | 237<br><br>246 |
| 4-propylnonane<br>$C_{12}H_{26}$<br>[6165-37-3]<br>QLKPGWNXMMBQMG-UHFFFAOYSA-N | $1.1\times10^{-6}$<br>$8.7\times10^{-7}$<br>$1.3\times10^{-6}$ | | Yaws (2003)<br>Gharagheizi et al. (2012)<br>Gharagheizi et al. (2010) | X<br>Q<br>Q | 237<br><br>246 |
| 5-propylnonane<br>$C_{12}H_{26}$<br>[998-35-6]<br>ISHSSTRAYNPQFX-UHFFFAOYSA-N | $1.1\times10^{-6}$<br>$8.2\times10^{-7}$<br>$1.3\times10^{-6}$ | | Yaws (2003)<br>Gharagheizi et al. (2012)<br>Gharagheizi et al. (2010) | X<br>Q<br>Q | 237<br><br>246 |
| 4-isopropylnonane<br>$C_{12}H_{26}$<br>[62184-71-8]<br>BXYKZQOGHMLDIE-UHFFFAOYSA-N | $1.0\times10^{-6}$<br>$8.5\times10^{-7}$<br>$1.0\times10^{-6}$ | | Yaws (2003)<br>Gharagheizi et al. (2012)<br>Gharagheizi et al. (2010) | X<br>Q<br>Q | 237<br><br>246 |
| 5-isopropylnonane<br>$C_{12}H_{26}$<br>[62184-72-9]<br>ALZCRHWYCQKCQK-UHFFFAOYSA-N | $1.0\times10^{-6}$<br>$8.5\times10^{-7}$<br>$1.0\times10^{-6}$ | | Yaws (2003)<br>Gharagheizi et al. (2012)<br>Gharagheizi et al. (2010) | X<br>Q<br>Q | 237<br><br>246 |
| 2,2,3,3-tetramethyloctane<br>$C_{12}H_{26}$<br>[62183-74-8]<br>UXQAEOWCSOPBLF-UHFFFAOYSA-N | $6.9\times10^{-7}$<br>$9.3\times10^{-7}$<br>$7.5\times10^{-7}$ | | Yaws (2003)<br>Gharagheizi et al. (2012)<br>Gharagheizi et al. (2010) | X<br>Q<br>Q | 237<br><br>246 |
| 2,2,3,4-tetramethyloctane<br>$C_{12}H_{26}$<br>[62183-75-9]<br>HEJULKHHKGYPNT-UHFFFAOYSA-N | $7.8\times10^{-7}$<br>$8.0\times10^{-7}$<br>$8.0\times10^{-7}$ | | Yaws (2003)<br>Gharagheizi et al. (2012)<br>Gharagheizi et al. (2010) | X<br>Q<br>Q | 237<br><br>246 |
| 2,2,3,5-tetramethyloctane<br>$C_{12}H_{26}$<br>[62183-76-0]<br>MRBTZFDZTJYCAO-UHFFFAOYSA-N | $8.0\times10^{-7}$<br>$7.3\times10^{-7}$<br>$8.0\times10^{-7}$ | | Yaws (2003)<br>Gharagheizi et al. (2012)<br>Gharagheizi et al. (2010) | X<br>Q<br>Q | 237<br><br>246 |





Table A2.1: Alkanes (...continued)

| Substance Formula (Trivial Name) [CAS Registry Number] InChIKey | $H_s^{cp}$ (at $T^\ominus$) $\left[\dfrac{\text{mol}}{\text{m}^3\,\text{Pa}}\right]$ | $\dfrac{\text{d}\ln H_s^{cp}}{\text{d}(1/T)}$ [K] | Reference | Type | Note |
|---|---|---|---|---|---|
| 2,2,3,6-tetramethyloctane C$_{12}$H$_{26}$ [62183-77-1] YRSBZJIBIXVOME-UHFFFAOYSA-N | $7.9\times10^{-7}$ $7.8\times10^{-7}$ $8.0\times10^{-7}$ | | Yaws (2003) Gharagheizi et al. (2012) Gharagheizi et al. (2010) | X Q Q | 237 246 |
| 2,2,3,7-tetramethyloctane C$_{12}$H$_{26}$ [62183-78-2] CPEHPCZXBBEYOL-UHFFFAOYSA-N | $7.9\times10^{-7}$ $7.8\times10^{-7}$ $8.0\times10^{-7}$ | | Yaws (2003) Gharagheizi et al. (2012) Gharagheizi et al. (2010) | X Q Q | 237 246 |
| 2,2,4,4-tetramethyloctane C$_{12}$H$_{26}$ [62183-79-3] TYUFTNSABIBNRY-UHFFFAOYSA-N | $7.2\times10^{-7}$ $7.3\times10^{-7}$ $7.5\times10^{-7}$ | | Yaws (2003) Gharagheizi et al. (2012) Gharagheizi et al. (2010) | X Q Q | 237 246 |
| 2,2,4,5-tetramethyloctane C$_{12}$H$_{26}$ [62183-80-6] SNJCVPBJFQJMFW-UHFFFAOYSA-N | $8.3\times10^{-7}$ $6.3\times10^{-7}$ $8.0\times10^{-7}$ | | Yaws (2003) Gharagheizi et al. (2012) Gharagheizi et al. (2010) | X Q Q | 237 246 |
| 2,2,4,6-tetramethyloctane C$_{12}$H$_{26}$ [62183-81-7] ODZRFEPMTHWMBK-UHFFFAOYSA-N | $8.4\times10^{-7}$ $6.0\times10^{-7}$ $8.0\times10^{-7}$ | | Yaws (2003) Gharagheizi et al. (2012) Gharagheizi et al. (2010) | X Q Q | 237 246 |
| 2,2,4,7-tetramethyloctane C$_{12}$H$_{26}$ [62183-82-8] QHDQKPZLRKHEHV-UHFFFAOYSA-N | $8.4\times10^{-7}$ $6.0\times10^{-7}$ $8.0\times10^{-7}$ | | Yaws (2003) Gharagheizi et al. (2012) Gharagheizi et al. (2010) | X Q Q | 237 246 |
| 2,2,5,5-tetramethyloctane C$_{12}$H$_{26}$ [62183-83-9] RJJMZKVKQLGGDK-UHFFFAOYSA-N | $7.8\times10^{-7}$ $5.8\times10^{-7}$ $7.5\times10^{-7}$ | | Yaws (2003) Gharagheizi et al. (2012) Gharagheizi et al. (2010) | X Q Q | 237 246 |
| 2,2,5,6-tetramethyloctane C$_{12}$H$_{26}$ [62183-84-0] BQUVPJLOJUCQGM-UHFFFAOYSA-N | $8.1\times10^{-7}$ $6.9\times10^{-7}$ $8.0\times10^{-7}$ | | Yaws (2003) Gharagheizi et al. (2012) Gharagheizi et al. (2010) | X Q Q | 237 246 |
| 2,2,5,7-tetramethyloctane C$_{12}$H$_{26}$ [62199-19-3] ISTINYWNHRBEDQ-UHFFFAOYSA-N | $8.4\times10^{-7}$ $6.0\times10^{-7}$ $8.0\times10^{-7}$ | | Yaws (2003) Gharagheizi et al. (2012) Gharagheizi et al. (2010) | X Q Q | 237 246 |
| 2,2,6,6-tetramethyloctane C$_{12}$H$_{26}$ [62199-20-6] BANZIZHQGFRTMI-UHFFFAOYSA-N | $7.7\times10^{-7}$ $6.0\times10^{-7}$ $7.5\times10^{-7}$ | | Yaws (2003) Gharagheizi et al. (2012) Gharagheizi et al. (2010) | X Q Q | 237 246 |



Table A2.1: Alkanes (... continued)

| Substance Formula (Trivial Name) [CAS Registry Number] InChIKey | $H_s^{cp}$ (at $T^\ominus$) $\left[\dfrac{\text{mol}}{\text{m}^3\,\text{Pa}}\right]$ | $\dfrac{\text{d}\ln H_s^{cp}}{\text{d}(1/T)}$ [K] | Reference | Type | Note |
|---|---|---|---|---|---|
| 2,2,6,7-tetramethyloctane | $8.0\times10^{-7}$ | | Yaws (2003) | X | 237 |
| $C_{12}H_{26}$ | $7.1\times10^{-7}$ | | Gharagheizi et al. (2012) | Q | |
| [62199-21-7] | $8.0\times10^{-7}$ | | Gharagheizi et al. (2010) | Q | 246 |
| ZSXJNQURTCJFGP-UHFFFAOYSA-N | | | | | |
| 2,2,7,7-tetramethyloctane | $7.7\times10^{-7}$ | | Yaws (2003) | X | 237 |
| $C_{12}H_{26}$ | $6.0\times10^{-7}$ | | Gharagheizi et al. (2012) | Q | |
| [1071-31-4] | $7.5\times10^{-7}$ | | Gharagheizi et al. (2010) | Q | 246 |
| QZUFNKONEPLWBC-UHFFFAOYSA-N | | | | | |
| 2,3,3,4-tetramethyloctane | $7.6\times10^{-7}$ | | Yaws (2003) | X | 237 |
| $C_{12}H_{26}$ | $9.5\times10^{-7}$ | | Gharagheizi et al. (2012) | Q | |
| [62199-22-8] | $8.0\times10^{-7}$ | | Gharagheizi et al. (2010) | Q | 246 |
| ITTRGXLKGDMYDX-UHFFFAOYSA-N | | | | | |
| 2,3,3,5-tetramethyloctane | $7.9\times10^{-7}$ | | Yaws (2003) | X | 237 |
| $C_{12}H_{26}$ | $7.8\times10^{-7}$ | | Gharagheizi et al. (2012) | Q | |
| [62199-23-9] | $8.0\times10^{-7}$ | | Gharagheizi et al. (2010) | Q | 246 |
| RVOIBAZQSHFKMM-UHFFFAOYSA-N | | | | | |
| 2,3,3,6-tetramethyloctane | $7.8\times10^{-7}$ | | Yaws (2003) | X | 237 |
| $C_{12}H_{26}$ | $8.2\times10^{-7}$ | | Gharagheizi et al. (2012) | Q | |
| [62199-24-0] | $8.0\times10^{-7}$ | | Gharagheizi et al. (2010) | Q | 246 |
| MBJHOSHYIFXEPL-UHFFFAOYSA-N | | | | | |
| 2,3,3,7-tetramethyloctane | $7.8\times10^{-7}$ | | Yaws (2003) | X | 237 |
| $C_{12}H_{26}$ | $8.2\times10^{-7}$ | | Gharagheizi et al. (2012) | Q | |
| [62199-25-1] | $8.0\times10^{-7}$ | | Gharagheizi et al. (2010) | Q | 246 |
| WWQBXZREQWIXFF-UHFFFAOYSA-N | | | | | |
| 2,3,4,4-tetramethyloctane | $7.8\times10^{-7}$ | | Yaws (2003) | X | 237 |
| $C_{12}H_{26}$ | $8.2\times10^{-7}$ | | Gharagheizi et al. (2012) | Q | |
| [62199-26-2] | $8.0\times10^{-7}$ | | Gharagheizi et al. (2010) | Q | 246 |
| MELCVUZPHQRNTK-UHFFFAOYSA-N | | | | | |
| 2,3,4,5-tetramethyloctane | $8.5\times10^{-7}$ | | Yaws (2003) | X | 237 |
| $C_{12}H_{26}$ | $8.4\times10^{-7}$ | | Gharagheizi et al. (2012) | Q | |
| [62199-27-3] | $8.9\times10^{-7}$ | | Gharagheizi et al. (2010) | Q | 246 |
| AMIZVDCCXDQOHL-UHFFFAOYSA-N | | | | | |
| 2,3,4,6-tetramethyloctane | $8.6\times10^{-7}$ | | Yaws (2003) | X | 237 |
| $C_{12}H_{26}$ | $8.0\times10^{-7}$ | | Gharagheizi et al. (2012) | Q | |
| [62199-28-4] | $8.9\times10^{-7}$ | | Gharagheizi et al. (2010) | Q | 246 |
| OYZWSGWTBSWCLN-UHFFFAOYSA-N | | | | | |
| 2,3,4,7-tetramethyloctane | $8.7\times10^{-7}$ | | Yaws (2003) | X | 237 |
| $C_{12}H_{26}$ | $7.7\times10^{-7}$ | | Gharagheizi et al. (2012) | Q | |
| [62199-29-5] | $8.9\times10^{-7}$ | | Gharagheizi et al. (2010) | Q | 246 |
| YUCNZDFGMLKYFT-UHFFFAOYSA-N | | | | | |



Table A2.1: Alkanes (... continued)

| Substance<br>Formula<br>(Trivial Name)<br>[CAS Registry Number]<br>InChIKey | $H_s^{cp}$<br>(at $T^{\ominus}$)<br>$\left[\dfrac{\mathrm{mol}}{\mathrm{m}^3\,\mathrm{Pa}}\right]$ | $\dfrac{\mathrm{d}\ln H_s^{cp}}{\mathrm{d}(1/T)}$<br><br>[K] | Reference | Type | Note |
|---|---|---|---|---|---|
| 2,3,5,5-tetramethyloctane<br>$C_{12}H_{26}$<br>[62199-30-8]<br>RQOLUXQWYAKKTG-UHFFFAOYSA-N | $7.8\times10^{-7}$<br>$8.0\times10^{-7}$<br>$8.0\times10^{-7}$ | | Yaws (2003)<br>Gharagheizi et al. (2012)<br>Gharagheizi et al. (2010) | X<br>Q<br>Q | 237<br><br>246 |
| 2,3,5,6-tetramethyloctane<br>$C_{12}H_{26}$<br>[62199-31-9]<br>IPYLOMHOPWIUFA-UHFFFAOYSA-N | $8.6\times10^{-7}$<br>$8.0\times10^{-7}$<br>$8.9\times10^{-7}$ | | Yaws (2003)<br>Gharagheizi et al. (2012)<br>Gharagheizi et al. (2010) | X<br>Q<br>Q | 237<br><br>246 |
| 2,3,5,7-tetramethyloctane<br>$C_{12}H_{26}$<br>[62199-32-0]<br>AIUMUUQTYGQPMV-UHFFFAOYSA-N | $8.9\times10^{-7}$<br>$6.9\times10^{-7}$<br>$8.9\times10^{-7}$ | | Yaws (2003)<br>Gharagheizi et al. (2012)<br>Gharagheizi et al. (2010) | X<br>Q<br>Q | 237<br><br>246 |
| 2,3,6,6-tetramethyloctane<br>$C_{12}H_{26}$<br>[62199-33-1]<br>CVRMZIOMFWYAPV-UHFFFAOYSA-N | $7.9\times10^{-7}$<br>$7.5\times10^{-7}$<br>$8.0\times10^{-7}$ | | Yaws (2003)<br>Gharagheizi et al. (2012)<br>Gharagheizi et al. (2010) | X<br>Q<br>Q | 237<br><br>246 |
| 2,3,6,7-tetramethyloctane<br>$C_{12}H_{26}$<br>[52670-34-5]<br>FZCGYGCYZRXLDY-UHFFFAOYSA-N | $8.6\times10^{-7}$<br>$8.2\times10^{-7}$<br>$8.9\times10^{-7}$ | | Yaws (2003)<br>Gharagheizi et al. (2012)<br>Gharagheizi et al. (2010) | X<br>Q<br>Q | 237<br><br>246 |
| 2,4,4,5-tetramethyloctane<br>$C_{12}H_{26}$<br>[62199-34-2]<br>OBWRJGVWBWXXAO-UHFFFAOYSA-N | $8.0\times10^{-7}$<br>$7.1\times10^{-7}$<br>$8.0\times10^{-7}$ | | Yaws (2003)<br>Gharagheizi et al. (2012)<br>Gharagheizi et al. (2010) | X<br>Q<br>Q | 237<br><br>246 |
| 2,4,4,6-tetramethyloctane<br>$C_{12}H_{26}$<br>[62199-35-3]<br>UQCGTRPILZWACX-UHFFFAOYSA-N | $8.4\times10^{-7}$<br>$6.0\times10^{-7}$<br>$8.0\times10^{-7}$ | | Yaws (2003)<br>Gharagheizi et al. (2012)<br>Gharagheizi et al. (2010) | X<br>Q<br>Q | 237<br><br>246 |
| 2,4,4,7-tetramethyloctane<br>$C_{12}H_{26}$<br>[35866-96-7]<br>QYUWMFZODNLLRJ-UHFFFAOYSA-N | $8.5\times10^{-7}$<br>$5.8\times10^{-7}$<br>$8.0\times10^{-7}$ | | Yaws (2003)<br>Gharagheizi et al. (2012)<br>Gharagheizi et al. (2010) | X<br>Q<br>Q | 237<br><br>246 |
| 2,4,5,5-tetramethyloctane<br>$C_{12}H_{26}$<br>[62199-36-4]<br>OMFWIGPPTXTQHH-UHFFFAOYSA-N | $8.0\times10^{-7}$<br>$7.3\times10^{-7}$<br>$8.0\times10^{-7}$ | | Yaws (2003)<br>Gharagheizi et al. (2012)<br>Gharagheizi et al. (2010) | X<br>Q<br>Q | 237<br><br>246 |
| 2,4,5,6-tetramethyloctane<br>$C_{12}H_{26}$<br>[62199-37-5]<br>YJMGKHXCSVEVKG-UHFFFAOYSA-N | $8.7\times10^{-7}$<br>$7.7\times10^{-7}$<br>$8.9\times10^{-7}$ | | Yaws (2003)<br>Gharagheizi et al. (2012)<br>Gharagheizi et al. (2010) | X<br>Q<br>Q | 237<br><br>246 |





Table A2.1: Alkanes (. . . continued)

| Substance Formula (Trivial Name) [CAS Registry Number] InChIKey | $H_s^{cp}$ (at $T^{\ominus}$) $\left[\dfrac{\text{mol}}{\text{m}^3\,\text{Pa}}\right]$ | $\dfrac{\text{d}\ln H_s^{cp}}{\text{d}(1/T)}$ [K] | Reference | Type | Note |
|---|---|---|---|---|---|
| 2,4,5,7-tetramethyloctane $C_{12}H_{26}$ [2217-17-6] PWYLDJLMYGOPSR-UHFFFAOYSA-N | $9.1\times10^{-7}$ $6.2\times10^{-7}$ $8.9\times10^{-7}$ | | Yaws (2003) Gharagheizi et al. (2012) Gharagheizi et al. (2010) | X Q Q | 237 246 |
| 2,4,6,6-tetramethyloctane $C_{12}H_{26}$ [62199-38-6] VVEPRQWGSASPPP-UHFFFAOYSA-N | $8.3\times10^{-7}$ $6.3\times10^{-7}$ $8.0\times10^{-7}$ | | Yaws (2003) Gharagheizi et al. (2012) Gharagheizi et al. (2010) | X Q Q | 237 246 |
| 2,5,5,6-tetramethyloctane $C_{12}H_{26}$ [62199-39-7] DLCPMTDXNVOOLU-UHFFFAOYSA-N | $7.9\times10^{-7}$ $7.8\times10^{-7}$ $8.0\times10^{-7}$ | | Yaws (2003) Gharagheizi et al. (2012) Gharagheizi et al. (2010) | X Q Q | 237 246 |
| 2,5,6,6-tetramethyloctane $C_{12}H_{26}$ [62199-40-0] SQOVOQLOSFZJGI-UHFFFAOYSA-N | $7.8\times10^{-7}$ $8.2\times10^{-7}$ $8.0\times10^{-7}$ | | Yaws (2003) Gharagheizi et al. (2012) Gharagheizi et al. (2010) | X Q Q | 237 246 |
| 3,3,4,4-tetramethyloctane $C_{12}H_{26}$ [62199-41-1] VXEUNQVVVNDDTL-UHFFFAOYSA-N | $6.8\times10^{-7}$ $9.8\times10^{-7}$ $7.5\times10^{-7}$ | | Yaws (2003) Gharagheizi et al. (2012) Gharagheizi et al. (2010) | X Q Q | 237 246 |
| 3,3,4,5-tetramethyloctane $C_{12}H_{26}$ [62199-42-2] FATMBSLXTUXZBP-UHFFFAOYSA-N | $7.7\times10^{-7}$ $8.7\times10^{-7}$ $8.0\times10^{-7}$ | | Yaws (2003) Gharagheizi et al. (2012) Gharagheizi et al. (2010) | X Q Q | 237 246 |
| 3,3,4,6-tetramethyloctane $C_{12}H_{26}$ [62199-43-3] VCXMKRQQDPQALS-UHFFFAOYSA-N | $7.8\times10^{-7}$ $8.2\times10^{-7}$ $8.0\times10^{-7}$ | | Yaws (2003) Gharagheizi et al. (2012) Gharagheizi et al. (2010) | X Q Q | 237 246 |
| 3,3,5,5-tetramethyloctane $C_{12}H_{26}$ [62199-44-4] UJLDJWNVZROTLS-UHFFFAOYSA-N | $7.0\times10^{-7}$ $8.5\times10^{-7}$ $7.5\times10^{-7}$ | | Yaws (2003) Gharagheizi et al. (2012) Gharagheizi et al. (2010) | X Q Q | 237 246 |
| 3,3,5,6-tetramethyloctane $C_{12}H_{26}$ [62199-45-5] AQPSWWWUVHSZJK-UHFFFAOYSA-N | $7.9\times10^{-7}$ $7.5\times10^{-7}$ $8.0\times10^{-7}$ | | Yaws (2003) Gharagheizi et al. (2012) Gharagheizi et al. (2010) | X Q Q | 237 246 |
| 3,3,6,6-tetramethyloctane $C_{12}H_{26}$ [62199-46-6] PAEUGKMMSQUAGH-UHFFFAOYSA-N | $7.4\times10^{-7}$ $6.9\times10^{-7}$ $7.5\times10^{-7}$ | | Yaws (2003) Gharagheizi et al. (2012) Gharagheizi et al. (2010) | X Q Q | 237 246 |





Table A2.1: Alkanes (...continued)

| Substance Formula (Trivial Name) [CAS Registry Number] InChIKey | $H_s^{cp}$ (at $T^{\ominus}$) $\left[\dfrac{\text{mol}}{\text{m}^3\,\text{Pa}}\right]$ | $\dfrac{\text{d}\ln H_s^{cp}}{\text{d}(1/T)}$ [K] | Reference | Type | Note |
|---|---|---|---|---|---|
| 3,4,4,5-tetramethyloctane $C_{12}H_{26}$ [62199-47-7] HWNTXBYEGXKWKG-UHFFFAOYSA-N | $7.6\times10^{-7}$ $9.5\times10^{-7}$ $8.0\times10^{-7}$ | | Yaws (2003) Gharagheizi et al. (2012) Gharagheizi et al. (2010) | X Q Q | 237 246 |
| 3,4,4,6-tetramethyloctane $C_{12}H_{26}$ [62185-19-7] LNRFUGUGPGLBLV-UHFFFAOYSA-N | $7.8\times10^{-7}$ $8.0\times10^{-7}$ $8.0\times10^{-7}$ | | Yaws (2003) Gharagheizi et al. (2012) Gharagheizi et al. (2010) | X Q Q | 237 246 |
| 3,4,5,5-tetramethyloctane $C_{12}H_{26}$ [62185-20-0] NCNZUEBQNGRARF-UHFFFAOYSA-N | $7.7\times10^{-7}$ $8.5\times10^{-7}$ $8.0\times10^{-7}$ | | Yaws (2003) Gharagheizi et al. (2012) Gharagheizi et al. (2010) | X Q Q | 237 246 |
| 3,4,5,6-tetramethyloctane $C_{12}H_{26}$ [62185-21-1] NADJQGPTQSFIHB-UHFFFAOYSA-N | $8.5\times10^{-7}$ $8.9\times10^{-7}$ $8.9\times10^{-7}$ | | Yaws (2003) Gharagheizi et al. (2012) Gharagheizi et al. (2010) | X Q Q | 237 246 |
| 4,4,5,5-tetramethyloctane $C_{12}H_{26}$ [62185-22-2] INTYEXUWOYUVGJ-UHFFFAOYSA-N | $6.9\times10^{-7}$ $9.3\times10^{-7}$ $7.5\times10^{-7}$ | | Yaws (2003) Gharagheizi et al. (2012) Gharagheizi et al. (2010) | X Q Q | 237 246 |
| 2,2-dimethyl-3-ethyloctane $C_{12}H_{26}$ [62183-95-3] LIZQPVRBTUZZCL-UHFFFAOYSA-N | $8.5\times10^{-7}$ $8.2\times10^{-7}$ $8.7\times10^{-7}$ | | Yaws (2003) Gharagheizi et al. (2012) Gharagheizi et al. (2010) | X Q Q | 237 246 |
| 2,2-dimethyl-4-ethyloctane $C_{12}H_{26}$ [62183-96-4] PEGJIJFSOTZJFH-UHFFFAOYSA-N | $9.0\times10^{-7}$ $6.3\times10^{-7}$ $8.7\times10^{-7}$ | | Yaws (2003) Gharagheizi et al. (2012) Gharagheizi et al. (2010) | X Q Q | 237 246 |
| 2,2-dimethyl-5-ethyloctane $C_{12}H_{26}$ [62183-97-5] ACWGGQUVVQOZPJ-UHFFFAOYSA-N | $8.9\times10^{-7}$ $6.7\times10^{-7}$ $8.7\times10^{-7}$ | | Yaws (2003) Gharagheizi et al. (2012) Gharagheizi et al. (2010) | X Q Q | 237 246 |
| 2,2-dimethyl-6-ethyloctane $C_{12}H_{26}$ [62183-98-6] SHNILIMSRTWRKG-UHFFFAOYSA-N | $8.6\times10^{-7}$ $7.5\times10^{-7}$ $8.7\times10^{-7}$ | | Yaws (2003) Gharagheizi et al. (2012) Gharagheizi et al. (2010) | X Q Q | 237 246 |
| 2,3-dimethyl-3-ethyloctane $C_{12}H_{26}$ [62183-99-7] NDFHPGVSCVJNFR-UHFFFAOYSA-N | $8.3\times10^{-7}$ $9.7\times10^{-7}$ $8.7\times10^{-7}$ | | Yaws (2003) Gharagheizi et al. (2012) Gharagheizi et al. (2010) | X Q Q | 237 246 |





Table A2.1: Alkanes (...continued)

| Substance Formula (Trivial Name) [CAS Registry Number] InChIKey | $H_s^{cp}$ (at $T^{\ominus}$) $\left[\dfrac{\text{mol}}{\text{m}^3\,\text{Pa}}\right]$ | $\dfrac{\text{d}\ln H_s^{cp}}{\text{d}(1/T)}$ [K] | Reference | Type | Note |
|---|---|---|---|---|---|
| 2,3-dimethyl-4-ethyloctane | $9.3\times10^{-7}$ | | Yaws (2003) | X | 237 |
| $C_{12}H_{26}$ | $8.4\times10^{-7}$ | | Gharagheizi et al. (2012) | Q | |
| [62184-00-3] | $9.2\times10^{-7}$ | | Gharagheizi et al. (2010) | Q | 246 |
| DCBIYDZULXWXTQ-UHFFFAOYSA-N | | | | | |
| 2,3-dimethyl-5-ethyloctane | $9.5\times10^{-7}$ | | Yaws (2003) | X | 237 |
| $C_{12}H_{26}$ | $7.7\times10^{-7}$ | | Gharagheizi et al. (2012) | Q | |
| [62184-01-4] | $9.2\times10^{-7}$ | | Gharagheizi et al. (2010) | Q | 246 |
| NPUNZBBVZZNRNS-UHFFFAOYSA-N | | | | | |
| 2,3-dimethyl-6-ethyloctane | $9.3\times10^{-7}$ | | Yaws (2003) | X | 237 |
| $C_{12}H_{26}$ | $8.9\times10^{-7}$ | | Gharagheizi et al. (2012) | Q | |
| [62184-02-5] | $9.2\times10^{-7}$ | | Gharagheizi et al. (2010) | Q | 246 |
| BWSXPPUMTZVKPA-UHFFFAOYSA-N | | | | | |
| 2,4-dimethyl-3-ethyloctane | $9.3\times10^{-7}$ | | Yaws (2003) | X | 237 |
| $C_{12}H_{26}$ | $8.4\times10^{-7}$ | | Gharagheizi et al. (2012) | Q | |
| [62184-03-6] | $9.2\times10^{-7}$ | | Gharagheizi et al. (2010) | Q | 246 |
| QYHDQDAUJCPVFJ-UHFFFAOYSA-N | | | | | |
| 2,4-dimethyl-4-ethyloctane | $8.8\times10^{-7}$ | | Yaws (2003) | X | 237 |
| $C_{12}H_{26}$ | $7.1\times10^{-7}$ | | Gharagheizi et al. (2012) | Q | |
| [62184-04-7] | $8.7\times10^{-7}$ | | Gharagheizi et al. (2010) | Q | 246 |
| XAKSPVTVTMAMDT-UHFFFAOYSA-N | | | | | |
| 2,4-dimethyl-5-ethyloctane | $9.5\times10^{-7}$ | | Yaws (2003) | X | 237 |
| $C_{12}H_{26}$ | $7.5\times10^{-7}$ | | Gharagheizi et al. (2012) | Q | |
| [62184-05-8] | $9.2\times10^{-7}$ | | Gharagheizi et al. (2010) | Q | 246 |
| JHDZOLPKXSWZKD-UHFFFAOYSA-N | | | | | |
| 2,4-dimethyl-6-ethyloctane | $9.6\times10^{-7}$ | | Yaws (2003) | X | 237 |
| $C_{12}H_{26}$ | $7.2\times10^{-7}$ | | Gharagheizi et al. (2012) | Q | |
| [62184-06-9] | $9.2\times10^{-7}$ | | Gharagheizi et al. (2010) | Q | 246 |
| VGDSKYYXZZCASQ-UHFFFAOYSA-N | | | | | |
| 2,5-dimethyl-3-ethyloctane | $9.5\times10^{-7}$ | | Yaws (2003) | X | 237 |
| $C_{12}H_{26}$ | $7.7\times10^{-7}$ | | Gharagheizi et al. (2012) | Q | |
| [62184-07-0] | $9.2\times10^{-7}$ | | Gharagheizi et al. (2010) | Q | 246 |
| BCQWMBQACXNHGP-UHFFFAOYSA-N | | | | | |
| 2,5-dimethyl-4-ethyloctane | $9.6\times10^{-7}$ | | Yaws (2003) | X | 237 |
| $C_{12}H_{26}$ | $7.2\times10^{-7}$ | | Gharagheizi et al. (2012) | Q | |
| [62184-08-1] | $9.2\times10^{-7}$ | | Gharagheizi et al. (2010) | Q | 246 |
| IMMDGMKEESKIQT-UHFFFAOYSA-N | | | | | |
| 2,5-dimethyl-5-ethyloctane | $8.6\times10^{-7}$ | | Yaws (2003) | X | 237 |
| $C_{12}H_{26}$ | $7.5\times10^{-7}$ | | Gharagheizi et al. (2012) | Q | |
| [62184-09-2] | $8.7\times10^{-7}$ | | Gharagheizi et al. (2010) | Q | 246 |
| BVAIFAWSJOPLJD-UHFFFAOYSA-N | | | | | |



Table A2.1: Alkanes (...continued)

| Substance Formula (Trivial Name) [CAS Registry Number] InChIKey | $H_s^{cp}$ (at $T^{\ominus}$) $\left[\dfrac{\text{mol}}{\text{m}^3\,\text{Pa}}\right]$ | $\dfrac{\text{d}\ln H_s^{cp}}{\text{d}(1/T)}$ [K] | Reference | Type | Note |
|---|---|---|---|---|---|
| 2,5-dimethyl-6-ethyloctane | $9.3\times10^{-7}$ | | Yaws (2003) | X | 237 |
| $C_{12}H_{26}$ | $8.4\times10^{-7}$ | | Gharagheizi et al. (2012) | Q | |
| [62183-50-0] | $9.2\times10^{-7}$ | | Gharagheizi et al. (2010) | Q | 246 |
| CYTDZGRTFHXLQS-UHFFFAOYSA-N | | | | | |
| 2,6-dimethyl-3-ethyloctane | $9.3\times10^{-7}$ | | Yaws (2003) | X | 237 |
| $C_{12}H_{26}$ | $8.4\times10^{-7}$ | | Gharagheizi et al. (2012) | Q | |
| [62183-51-1] | $9.2\times10^{-7}$ | | Gharagheizi et al. (2010) | Q | 246 |
| WAFCPUFGSNTFSP-UHFFFAOYSA-N | | | | | |
| 2,6-dimethyl-4-ethyloctane | $9.7\times10^{-7}$ | | Yaws (2003) | X | 237 |
| $C_{12}H_{26}$ | $6.8\times10^{-7}$ | | Gharagheizi et al. (2012) | Q | |
| [62183-52-2] | $9.2\times10^{-7}$ | | Gharagheizi et al. (2010) | Q | 246 |
| JIOQMQLJPQDNFA-UHFFFAOYSA-N | | | | | |
| 2,6-dimethyl-5-ethyloctane | $9.4\times10^{-7}$ | | Yaws (2003) | X | 237 |
| $C_{12}H_{26}$ | $7.9\times10^{-7}$ | | Gharagheizi et al. (2012) | Q | |
| [62183-53-3] | $9.2\times10^{-7}$ | | Gharagheizi et al. (2010) | Q | 246 |
| ZBMWKURVBLDBAH-UHFFFAOYSA-N | | | | | |
| 2,6-dimethyl-6-ethyloctane | $8.4\times10^{-7}$ | | Yaws (2003) | X | 237 |
| $C_{12}H_{26}$ | $8.6\times10^{-7}$ | | Gharagheizi et al. (2012) | Q | |
| [62183-54-4] | $8.7\times10^{-7}$ | | Gharagheizi et al. (2010) | Q | 246 |
| DGFWFJRTTJTVEQ-UHFFFAOYSA-N | | | | | |
| 2,7-dimethyl-3-ethyloctane | $9.4\times10^{-7}$ | | Yaws (2003) | X | 237 |
| $C_{12}H_{26}$ | $8.1\times10^{-7}$ | | Gharagheizi et al. (2012) | Q | |
| [62183-55-5] | $9.2\times10^{-7}$ | | Gharagheizi et al. (2010) | Q | 246 |
| XEMFRSYZKNPRTA-UHFFFAOYSA-N | | | | | |
| 2,7-dimethyl-4-ethyloctane | $9.7\times10^{-7}$ | | Yaws (2003) | X | 237 |
| $C_{12}H_{26}$ | $6.8\times10^{-7}$ | | Gharagheizi et al. (2012) | Q | |
| [62183-56-6] | $9.2\times10^{-7}$ | | Gharagheizi et al. (2010) | Q | 246 |
| KSVMIUVYLYZDMT-UHFFFAOYSA-N | | | | | |
| 3,3-dimethyl-4-ethyloctane | $8.4\times10^{-7}$ | | Yaws (2003) | X | 237 |
| $C_{12}H_{26}$ | $8.6\times10^{-7}$ | | Gharagheizi et al. (2012) | Q | |
| [62183-57-7] | $8.7\times10^{-7}$ | | Gharagheizi et al. (2010) | Q | 246 |
| BKNJDUXGYCILJT-UHFFFAOYSA-N | | | | | |
| 3,3-dimethyl-5-ethyloctane | $8.8\times10^{-7}$ | | Yaws (2003) | X | 237 |
| $C_{12}H_{26}$ | $7.1\times10^{-7}$ | | Gharagheizi et al. (2012) | Q | |
| [62183-58-8] | $8.7\times10^{-7}$ | | Gharagheizi et al. (2010) | Q | 246 |
| GQTCKTISADWVNV-UHFFFAOYSA-N | | | | | |
| 3,3-dimethyl-6-ethyloctane | $8.5\times10^{-7}$ | | Yaws (2003) | X | 237 |
| $C_{12}H_{26}$ | $8.2\times10^{-7}$ | | Gharagheizi et al. (2012) | Q | |
| [62183-59-9] | $8.7\times10^{-7}$ | | Gharagheizi et al. (2010) | Q | 246 |
| DDRNTRVXHMVVLA-UHFFFAOYSA-N | | | | | |





Table A2.1: Alkanes (. . . continued)

| Substance Formula (Trivial Name) [CAS Registry Number] InChIKey | $H_s^{cp}$ (at $T^{\ominus}$) $\left[\dfrac{\mathrm{mol}}{\mathrm{m^3\,Pa}}\right]$ | $\dfrac{\mathrm{d}\ln H_s^{cp}}{\mathrm{d}(1/T)}$ [K] | Reference | Type | Note |
|---|---|---|---|---|---|
| 3,4-dimethyl-3-ethyloctane $C_{12}H_{26}$ [62212-28-6] AOQVLRSMPNHJPF-UHFFFAOYSA-N | $8.3\times10^{-7}$ $9.2\times10^{-7}$ $8.7\times10^{-7}$ | | Yaws (2003) Gharagheizi et al. (2012) Gharagheizi et al. (2010) | X Q Q | 237 246 |
| 3,4-dimethyl-4-ethyloctane $C_{12}H_{26}$ [62183-60-2] WBKPODGAYHVUMM-UHFFFAOYSA-N | $8.3\times10^{-7}$ $9.4\times10^{-7}$ $8.7\times10^{-7}$ | | Yaws (2003) Gharagheizi et al. (2012) Gharagheizi et al. (2010) | X Q Q | 237 246 |
| 3,4-dimethyl-5-ethyloctane $C_{12}H_{26}$ [62183-61-3] LEPWFUNEXVKPDO-UHFFFAOYSA-N | $9.3\times10^{-7}$ $8.6\times10^{-7}$ $9.2\times10^{-7}$ | | Yaws (2003) Gharagheizi et al. (2012) Gharagheizi et al. (2010) | X Q Q | 237 246 |
| 3,4-dimethyl-6-ethyloctane $C_{12}H_{26}$ [62183-62-4] DPDGQUDZYGOBAM-UHFFFAOYSA-N | $9.3\times10^{-7}$ $8.6\times10^{-7}$ $9.2\times10^{-7}$ | | Yaws (2003) Gharagheizi et al. (2012) Gharagheizi et al. (2010) | X Q Q | 237 246 |
| 3,5-dimethyl-3-ethyloctane $C_{12}H_{26}$ [62183-63-5] QYJKERUDQYOWRC-UHFFFAOYSA-N | $8.5\times10^{-7}$ $8.2\times10^{-7}$ $8.7\times10^{-7}$ | | Yaws (2003) Gharagheizi et al. (2012) Gharagheizi et al. (2010) | X Q Q | 237 246 |
| 3,5-dimethyl-4-ethyloctane $C_{12}H_{26}$ [62183-64-6] OMAJEXDENFMCIN-UHFFFAOYSA-N | $9.3\times10^{-7}$ $8.4\times10^{-7}$ $9.2\times10^{-7}$ | | Yaws (2003) Gharagheizi et al. (2012) Gharagheizi et al. (2010) | X Q Q | 237 246 |
| 3,5-dimethyl-5-ethyloctane $C_{12}H_{26}$ [62183-65-7] OLBJDMSGRVKJMF-UHFFFAOYSA-N | $8.8\times10^{-7}$ $6.9\times10^{-7}$ $8.7\times10^{-7}$ | | Yaws (2003) Gharagheizi et al. (2012) Gharagheizi et al. (2010) | X Q Q | 237 246 |
| 3,5-dimethyl-6-ethyloctane $C_{12}H_{26}$ [62183-66-8] LLHGAWREVJAITE-UHFFFAOYSA-N | $9.3\times10^{-7}$ $8.4\times10^{-7}$ $9.2\times10^{-7}$ | | Yaws (2003) Gharagheizi et al. (2012) Gharagheizi et al. (2010) | X Q Q | 237 246 |
| 3,6-dimethyl-3-ethyloctane $C_{12}H_{26}$ [62183-67-9] MNXPIOFJFYQCKE-UHFFFAOYSA-N | $8.4\times10^{-7}$ $8.9\times10^{-7}$ $8.7\times10^{-7}$ | | Yaws (2003) Gharagheizi et al. (2012) Gharagheizi et al. (2010) | X Q Q | 237 246 |
| 3,6-dimethyl-4-ethyloctane $C_{12}H_{26}$ [62183-68-0] RTVRGBUFZDOPIH-UHFFFAOYSA-N | $9.4\times10^{-7}$ $8.1\times10^{-7}$ $9.2\times10^{-7}$ | | Yaws (2003) Gharagheizi et al. (2012) Gharagheizi et al. (2010) | X Q Q | 237 246 |





Table A2.1: Alkanes (...continued)

| Substance Formula (Trivial Name) [CAS Registry Number] InChIKey | $H_s^{cp}$ (at $T^\ominus$) $\left[\dfrac{\text{mol}}{\text{m}^3\,\text{Pa}}\right]$ | $\dfrac{\text{d}\ln H_s^{cp}}{\text{d}(1/T)}$ [K] | Reference | Type | Note |
|---|---|---|---|---|---|
| 4,4-dimethyl-3-ethyloctane $C_{12}H_{26}$ [62183-69-1] RHSANRLUOZFJDW-UHFFFAOYSA-N | $8.4\times10^{-7}$ $8.6\times10^{-7}$ $8.7\times10^{-7}$ | | Yaws (2003) Gharagheizi et al. (2012) Gharagheizi et al. (2010) | X Q Q | 237 246 |
| 4,4-dimethyl-5-ethyloctane $C_{12}H_{26}$ [62183-70-4] BETVNWPSZXTJLK-UHFFFAOYSA-N | $8.5\times10^{-7}$ $8.2\times10^{-7}$ $8.7\times10^{-7}$ | | Yaws (2003) Gharagheizi et al. (2012) Gharagheizi et al. (2010) | X Q Q | 237 246 |
| 4,4-dimethyl-6-ethyloctane $C_{12}H_{26}$ [62183-71-5] UGEWSKWSWHZABT-UHFFFAOYSA-N | $8.8\times10^{-7}$ $7.1\times10^{-7}$ $8.7\times10^{-7}$ | | Yaws (2003) Gharagheizi et al. (2012) Gharagheizi et al. (2010) | X Q Q | 237 246 |
| 4,5-dimethyl-3-ethyloctane $C_{12}H_{26}$ [62183-72-6] MNDPOEACQSQPCJ-UHFFFAOYSA-N | $9.3\times10^{-7}$ $8.9\times10^{-7}$ $9.2\times10^{-7}$ | | Yaws (2003) Gharagheizi et al. (2012) Gharagheizi et al. (2010) | X Q Q | 237 246 |
| 4,5-dimethyl-4-ethyloctane $C_{12}H_{26}$ [62183-73-7] MOEUNMMPAYEJOZ-UHFFFAOYSA-N | $8.3\times10^{-7}$ $9.2\times10^{-7}$ $8.7\times10^{-7}$ | | Yaws (2003) Gharagheizi et al. (2012) Gharagheizi et al. (2010) | X Q Q | 237 246 |
| 3,3-diethyloctane $C_{12}H_{26}$ [17302-40-8] DGJISSKLLWWXTG-UHFFFAOYSA-N | $9.1\times10^{-7}$ $1.0\times10^{-6}$ $1.0\times10^{-6}$ | | Yaws (2003) Gharagheizi et al. (2012) Gharagheizi et al. (2010) | X Q Q | 237 246 |
| 3,4-diethyloctane $C_{12}H_{26}$ [62183-92-0] HILZBOWVGZXYGQ-UHFFFAOYSA-N | $1.0\times10^{-6}$ $8.8\times10^{-7}$ $1.0\times10^{-6}$ | | Yaws (2003) Gharagheizi et al. (2012) Gharagheizi et al. (2010) | X Q Q | 237 246 |
| 3,5-diethyloctane $C_{12}H_{26}$ [62183-93-1] DSYIMNMBWBEOHY-UHFFFAOYSA-N | $1.0\times10^{-6}$ $8.3\times10^{-7}$ $1.0\times10^{-6}$ | | Yaws (2003) Gharagheizi et al. (2012) Gharagheizi et al. (2010) | X Q Q | 237 246 |
| 3,6-diethyloctane $C_{12}H_{26}$ [62183-94-2] UTCTYSTYZOAAOS-UHFFFAOYSA-N | $1.0\times10^{-6}$ $9.6\times10^{-7}$ $1.0\times10^{-6}$ | | Yaws (2003) Gharagheizi et al. (2012) Gharagheizi et al. (2010) | X Q Q | 237 246 |
| 4,4-diethyloctane $C_{12}H_{26}$ [17312-42-4] FTEQKVNFZNDQJG-UHFFFAOYSA-N | $9.2\times10^{-7}$ $9.1\times10^{-7}$ $1.0\times10^{-6}$ | | Yaws (2003) Gharagheizi et al. (2012) Gharagheizi et al. (2010) | X Q Q | 237 246 |





Table A2.1: Alkanes (... continued)

| Substance<br>Formula<br>(Trivial Name)<br>[CAS Registry Number]<br>InChIKey | $H_s^{cp}$ (at $T^{\ominus}$) $\left[\dfrac{\mathrm{mol}}{\mathrm{m^3\,Pa}}\right]$ | $\dfrac{\mathrm{d}\ln H_s^{cp}}{\mathrm{d}(1/T)}$ [K] | Reference | Type | Note |
|---|---|---|---|---|---|
| 4,5-diethyloctane<br>$C_{12}H_{26}$<br>[1636-41-5]<br>XRVCHXCONYJHLU-UHFFFAOYSA-N | $1.0\times10^{-6}$<br>$8.3\times10^{-7}$<br>$1.0\times10^{-6}$ | | Yaws (2003)<br>Gharagheizi et al. (2012)<br>Gharagheizi et al. (2010) | X<br>Q<br>Q | 237<br><br>246 |
| 2-methyl-4-propyloctane<br>$C_{12}H_{26}$<br>[62184-33-2]<br>FJPVBTBEYVNWCA-UHFFFAOYSA-N | $1.1\times10^{-6}$<br>$7.0\times10^{-7}$<br>$1.0\times10^{-6}$ | | Yaws (2003)<br>Gharagheizi et al. (2012)<br>Gharagheizi et al. (2010) | X<br>Q<br>Q | 237<br><br>246 |
| 2-methyl-5-propyloctane<br>$C_{12}H_{26}$<br>[62184-34-3]<br>NYBDGQCIJCAIJL-UHFFFAOYSA-N | $1.1\times10^{-6}$<br>$7.0\times10^{-7}$<br>$1.0\times10^{-6}$ | | Yaws (2003)<br>Gharagheizi et al. (2012)<br>Gharagheizi et al. (2010) | X<br>Q<br>Q | 237<br><br>246 |
| 3-methyl-4-propyloctane<br>$C_{12}H_{26}$<br>[62184-35-4]<br>CTXPEFTVFFFIBQ-UHFFFAOYSA-N | $1.0\times10^{-6}$<br>$8.3\times10^{-7}$<br>$1.0\times10^{-6}$ | | Yaws (2003)<br>Gharagheizi et al. (2012)<br>Gharagheizi et al. (2010) | X<br>Q<br>Q | 237<br><br>246 |
| 3-methyl-5-propyloctane<br>$C_{12}H_{26}$<br>[62184-36-5]<br>VOSHMCRSZWNTLJ-UHFFFAOYSA-N | $1.0\times10^{-6}$<br>$7.6\times10^{-7}$<br>$1.0\times10^{-6}$ | | Yaws (2003)<br>Gharagheizi et al. (2012)<br>Gharagheizi et al. (2010) | X<br>Q<br>Q | 237<br><br>246 |
| 4-methyl-4-propyloctane<br>$C_{12}H_{26}$<br>[17312-41-3]<br>RNKCZSBPRRHTOZ-UHFFFAOYSA-N | $9.3\times10^{-7}$<br>$7.9\times10^{-7}$<br>$1.0\times10^{-6}$ | | Yaws (2003)<br>Gharagheizi et al. (2012)<br>Gharagheizi et al. (2010) | X<br>Q<br>Q | 237<br><br>246 |
| 4-methyl-5-propyloctane<br>$C_{12}H_{26}$<br>[62183-85-1]<br>MVUWSCFPURGAJF-UHFFFAOYSA-N | $1.0\times10^{-6}$<br>$8.1\times10^{-7}$<br>$1.0\times10^{-6}$ | | Yaws (2003)<br>Gharagheizi et al. (2012)<br>Gharagheizi et al. (2010) | X<br>Q<br>Q | 237<br><br>246 |
| 2-methyl-3-isopropyloctane<br>$C_{12}H_{26}$<br>[13287-19-9]<br>KBIUKRVYBYJVSS-UHFFFAOYSA-N | $9.4\times10^{-7}$<br>$8.1\times10^{-7}$<br>$9.2\times10^{-7}$ | | Yaws (2003)<br>Gharagheizi et al. (2012)<br>Gharagheizi et al. (2010) | X<br>Q<br>Q | 237<br><br>246 |
| 2-methyl-4-isopropyloctane<br>$C_{12}H_{26}$<br>[62183-86-2]<br>NRRICFUUPCDSKW-UHFFFAOYSA-N | $9.7\times10^{-7}$<br>$6.8\times10^{-7}$<br>$9.2\times10^{-7}$ | | Yaws (2003)<br>Gharagheizi et al. (2012)<br>Gharagheizi et al. (2010) | X<br>Q<br>Q | 237<br><br>246 |
| 2-methyl-5-isopropyloctane<br>$C_{12}H_{26}$<br>[62183-87-3]<br>OPKVPKXVSDEXOZ-UHFFFAOYSA-N | $9.6\times10^{-7}$<br>$7.0\times10^{-7}$<br>$9.2\times10^{-7}$ | | Yaws (2003)<br>Gharagheizi et al. (2012)<br>Gharagheizi et al. (2010) | X<br>Q<br>Q | 237<br><br>246 |



Table A2.1: Alkanes (...continued)

| Substance Formula (Trivial Name) [CAS Registry Number] InChIKey | $H_s^{cp}$ (at $T^{\ominus}$) $\left[ \dfrac{\text{mol}}{\text{m}^3\,\text{Pa}} \right]$ | $\dfrac{\text{d}\ln H_s^{cp}}{\text{d}(1/T)}$ [K] | Reference | Type | Note |
|---|---|---|---|---|---|
| 3-methyl-4-isopropyloctane | $9.4\times10^{-7}$ | | Yaws (2003) | X | 237 |
| $C_{12}H_{26}$ | $7.9\times10^{-7}$ | | Gharagheizi et al. (2012) | Q | |
| [62183-88-4] | $9.2\times10^{-7}$ | | Gharagheizi et al. (2010) | Q | 246 |
| UXTHXSNAZBLVOD-UHFFFAOYSA-N | | | | | |
| 3-methyl-5-isopropyloctane | $9.6\times10^{-7}$ | | Yaws (2003) | X | 237 |
| $C_{12}H_{26}$ | $7.2\times10^{-7}$ | | Gharagheizi et al. (2012) | Q | |
| [62183-89-5] | $9.2\times10^{-7}$ | | Gharagheizi et al. (2010) | Q | 246 |
| VEJPAYNPRZYJNN-UHFFFAOYSA-N | | | | | |
| 4-methyl-4-isopropyloctane | $8.4\times10^{-7}$ | | Yaws (2003) | X | 237 |
| $C_{12}H_{26}$ | $8.6\times10^{-7}$ | | Gharagheizi et al. (2012) | Q | |
| [62183-90-8] | $8.7\times10^{-7}$ | | Gharagheizi et al. (2010) | Q | 246 |
| KCNISWXEVSJYRS-UHFFFAOYSA-N | | | | | |
| 4-methyl-5-isopropyloctane | $9.5\times10^{-7}$ | | Yaws (2003) | X | 237 |
| $C_{12}H_{26}$ | $7.7\times10^{-7}$ | | Gharagheizi et al. (2012) | Q | |
| [62183-91-9] | $9.2\times10^{-7}$ | | Gharagheizi et al. (2010) | Q | 246 |
| HEFVKUUZAXCMQX-UHFFFAOYSA-N | | | | | |
| 4-*tert*-butyloctane | $8.8\times10^{-7}$ | | Yaws (2003) | X | 237 |
| $C_{12}H_{26}$ | $7.1\times10^{-7}$ | | Gharagheizi et al. (2012) | Q | |
| [62184-32-1] | $8.7\times10^{-7}$ | | Gharagheizi et al. (2010) | Q | 246 |
| RWILNGDOCMDMJW-UHFFFAOYSA-N | | | | | |
| 2,2,4,6,6-pentamethylheptane | $7.7\times10^{-7}$ | | Yaws (2003) | X | 237 |
| $C_{12}H_{26}$ | $4.9\times10^{-7}$ | | Gharagheizi et al. (2012) | Q | |
| [13475-82-6] | $1.1\times10^{-6}$ | | Zhang et al. (2010) | Q | 287, 288 |
| VKPSKYDESGTTFR-UHFFFAOYSA-N | $2.3\times10^{-7}$ | | Zhang et al. (2010) | Q | 287, 289 |
| | $2.4\times10^{-5}$ | | Zhang et al. (2010) | Q | 287, 290 |
| | $5.1\times10^{-6}$ | | Zhang et al. (2010) | Q | 287, 291 |
| | $7.2\times10^{-7}$ | | Gharagheizi et al. (2010) | Q | 246 |
| 2,2,3,3,4-pentamethylheptane | $6.3\times10^{-7}$ | | Yaws (2003) | X | 237 |
| $C_{12}H_{26}$ | $9.3\times10^{-7}$ | | Gharagheizi et al. (2012) | Q | |
| [62198-80-5] | $7.2\times10^{-7}$ | | Gharagheizi et al. (2010) | Q | 246 |
| NTDYBHYVUCYILC-UHFFFAOYSA-N | | | | | |
| 2,2,3,3,5-pentamethylheptane | $6.5\times10^{-7}$ | | Yaws (2003) | X | 237 |
| $C_{12}H_{26}$ | $7.9\times10^{-7}$ | | Gharagheizi et al. (2012) | Q | |
| [62198-81-6] | $7.2\times10^{-7}$ | | Gharagheizi et al. (2010) | Q | 246 |
| ZLCPPJIFAFIYPV-UHFFFAOYSA-N | | | | | |
| 2,2,3,3,6-pentamethylheptane | $6.6\times10^{-7}$ | | Yaws (2003) | X | 237 |
| $C_{12}H_{26}$ | $7.6\times10^{-7}$ | | Gharagheizi et al. (2012) | Q | |
| [62198-82-7] | $7.2\times10^{-7}$ | | Gharagheizi et al. (2010) | Q | 246 |
| LYTFNNQFUUFUDH-UHFFFAOYSA-N | | | | | |



Table A2.1: Alkanes (. . . continued)

| Substance<br>Formula<br>(Trivial Name)<br>[CAS Registry Number]<br>InChIKey | $H_s^{cp}$<br>(at $T^{\ominus}$)<br>$\left[\dfrac{\mathrm{mol}}{\mathrm{m^3\,Pa}}\right]$ | $\dfrac{\mathrm{d}\ln H_s^{cp}}{\mathrm{d}(1/T)}$<br><br>[K] | Reference | Type | Note |
|---|---|---|---|---|---|
| 2,2,3,4,4-pentamethylheptane<br>$C_{12}H_{26}$<br>[62198-83-8]<br>SUFPUVGFEWPIFA-UHFFFAOYSA-N | $6.2\times10^{-7}$<br>$9.9\times10^{-7}$<br>$7.2\times10^{-7}$ | | Yaws (2003)<br>Gharagheizi et al. (2012)<br>Gharagheizi et al. (2010) | X<br>Q<br>Q | 237<br><br>246 |
| 2,2,3,4,5-pentamethylheptane<br>$C_{12}H_{26}$<br>[62198-84-9]<br>XYDYIKVIGCXDLV-UHFFFAOYSA-N | $7.2\times10^{-7}$<br>$7.8\times10^{-7}$<br>$8.0\times10^{-7}$ | | Yaws (2003)<br>Gharagheizi et al. (2012)<br>Gharagheizi et al. (2010) | X<br>Q<br>Q | 237<br><br>246 |
| 2,2,3,4,6-pentamethylheptane<br>$C_{12}H_{26}$<br>[62198-85-0]<br>XRJGBYMRUIWAJP-UHFFFAOYSA-N | $7.5\times10^{-7}$<br>$6.6\times10^{-7}$<br>$8.0\times10^{-7}$ | | Yaws (2003)<br>Gharagheizi et al. (2012)<br>Gharagheizi et al. (2010) | X<br>Q<br>Q | 237<br><br>246 |
| 2,2,3,5,5-pentamethylheptane<br>$C_{12}H_{26}$<br>[62198-86-1]<br>OSZLORKBJVWZNT-UHFFFAOYSA-N | $6.9\times10^{-7}$<br>$6.4\times10^{-7}$<br>$7.2\times10^{-7}$ | | Yaws (2003)<br>Gharagheizi et al. (2012)<br>Gharagheizi et al. (2010) | X<br>Q<br>Q | 237<br><br>246 |
| 2,2,3,5,6-pentamethylheptane<br>$C_{12}H_{26}$<br>[62198-87-2]<br>MHJUEZDXDYXCPA-UHFFFAOYSA-N | $7.5\times10^{-7}$<br>$6.7\times10^{-7}$<br>$8.0\times10^{-7}$ | | Yaws (2003)<br>Gharagheizi et al. (2012)<br>Gharagheizi et al. (2010) | X<br>Q<br>Q | 237<br><br>246 |
| 2,2,3,6,6-pentamethylheptane<br>$C_{12}H_{26}$<br>[62198-88-3]<br>SLGYOGBODYIQGW-UHFFFAOYSA-N | $7.1\times10^{-7}$<br>$6.1\times10^{-7}$<br>$7.2\times10^{-7}$ | | Yaws (2003)<br>Gharagheizi et al. (2012)<br>Gharagheizi et al. (2010) | X<br>Q<br>Q | 237<br><br>246 |
| 2,2,4,4,5-pentamethylheptane<br>$C_{12}H_{26}$<br>[62199-61-5]<br>XRGPEBQZXVFCRE-UHFFFAOYSA-N | $6.4\times10^{-7}$<br>$8.3\times10^{-7}$<br>$7.2\times10^{-7}$ | | Yaws (2003)<br>Gharagheizi et al. (2012)<br>Gharagheizi et al. (2010) | X<br>Q<br>Q | 237<br><br>246 |
| 2,2,4,4,6-pentamethylheptane<br>$C_{12}H_{26}$<br>[62199-62-6]<br>NOFQKTWPZFUCOO-UHFFFAOYSA-N | $7.0\times10^{-7}$<br>$6.2\times10^{-7}$<br>$7.2\times10^{-7}$ | | Yaws (2003)<br>Gharagheizi et al. (2012)<br>Gharagheizi et al. (2010) | X<br>Q<br>Q | 237<br><br>246 |
| 2,2,4,5,5-pentamethylheptane<br>$C_{12}H_{26}$<br>[62199-63-7]<br>JGOKXUSMFJOVQT-UHFFFAOYSA-N | $7.0\times10^{-7}$<br>$6.2\times10^{-7}$<br>$7.2\times10^{-7}$ | | Yaws (2003)<br>Gharagheizi et al. (2012)<br>Gharagheizi et al. (2010) | X<br>Q<br>Q | 237<br><br>246 |
| 2,2,4,5,6-pentamethylheptane<br>$C_{12}H_{26}$<br>[62199-64-8]<br>JVESFBBBMJIZLO-UHFFFAOYSA-N | $7.8\times10^{-7}$<br>$6.0\times10^{-7}$<br>$8.0\times10^{-7}$ | | Yaws (2003)<br>Gharagheizi et al. (2012)<br>Gharagheizi et al. (2010) | X<br>Q<br>Q | 237<br><br>246 |



Table A2.1: Alkanes (...continued)

| Substance Formula (Trivial Name) [CAS Registry Number] InChIKey | $H_s^{cp}$ (at $T^{\ominus}$) $\left[\dfrac{\mathrm{mol}}{\mathrm{m^3\,Pa}}\right]$ | $\dfrac{\mathrm{d}\ln H_s^{cp}}{\mathrm{d}(1/T)}$ [K] | Reference | Type | Note |
|---|---|---|---|---|---|
| 2,2,5,5,6-pentamethylheptane $C_{12}H_{26}$ [62199-65-9] FMJMMNPKKCFHTH-UHFFFAOYSA-N | $7.0\times10^{-7}$ $6.2\times10^{-7}$ $7.2\times10^{-7}$ | | Yaws (2003) Gharagheizi et al. (2012) Gharagheizi et al. (2010) | X Q Q | 237 246 |
| 2,3,3,4,4-pentamethylheptane $C_{12}H_{26}$ [62199-66-0] VBOBCKMZHQJDQA-UHFFFAOYSA-N | $6.2\times10^{-7}$ $1.0\times10^{-6}$ $7.2\times10^{-7}$ | | Yaws (2003) Gharagheizi et al. (2012) Gharagheizi et al. (2010) | X Q Q | 237 246 |
| 2,3,3,4,5-pentamethylheptane $C_{12}H_{26}$ [62199-67-1] OLZJEASBDLLWCY-UHFFFAOYSA-N | $7.0\times10^{-7}$ $9.3\times10^{-7}$ $8.0\times10^{-7}$ | | Yaws (2003) Gharagheizi et al. (2012) Gharagheizi et al. (2010) | X Q Q | 237 246 |
| 2,3,3,4,6-pentamethylheptane $C_{12}H_{26}$ [62199-68-2] YENAHMOHMFYKPC-UHFFFAOYSA-N | $7.2\times10^{-7}$ $8.1\times10^{-7}$ $8.0\times10^{-7}$ | | Yaws (2003) Gharagheizi et al. (2012) Gharagheizi et al. (2010) | X Q Q | 237 246 |
| 2,3,3,5,5-pentamethylheptane $C_{12}H_{26}$ [62199-69-3] DTLBRJVRUQMSQN-UHFFFAOYSA-N | $6.3\times10^{-7}$ $9.3\times10^{-7}$ $7.2\times10^{-7}$ | | Yaws (2003) Gharagheizi et al. (2012) Gharagheizi et al. (2010) | X Q Q | 237 246 |
| 2,3,3,5,6-pentamethylheptane $C_{12}H_{26}$ [52670-35-6] SZQDOCSTKXRXAR-UHFFFAOYSA-N | $7.3\times10^{-7}$ $7.4\times10^{-7}$ $8.0\times10^{-7}$ | | Yaws (2003) Gharagheizi et al. (2012) Gharagheizi et al. (2010) | X Q Q | 237 246 |
| 2,3,4,4,5-pentamethylheptane $C_{12}H_{26}$ [62199-70-6] CHHJLGDSEYTKKE-UHFFFAOYSA-N | $7.0\times10^{-7}$ $9.3\times10^{-7}$ $8.0\times10^{-7}$ | | Yaws (2003) Gharagheizi et al. (2012) Gharagheizi et al. (2010) | X Q Q | 237 246 |
| 2,3,4,4,6-pentamethylheptane $C_{12}H_{26}$ [62199-71-7] DHXWMADSONSWTL-UHFFFAOYSA-N | $7.4\times10^{-7}$ $7.0\times10^{-7}$ $8.0\times10^{-7}$ | | Yaws (2003) Gharagheizi et al. (2012) Gharagheizi et al. (2010) | X Q Q | 237 246 |
| 2,3,4,5,5-pentamethylheptane $C_{12}H_{26}$ [62199-72-8] BDFJISGXUJUYCX-UHFFFAOYSA-N | $7.1\times10^{-7}$ $8.5\times10^{-7}$ $8.0\times10^{-7}$ | | Yaws (2003) Gharagheizi et al. (2012) Gharagheizi et al. (2010) | X Q Q | 237 246 |
| 2,3,4,5,6-pentamethylheptane $C_{12}H_{26}$ [27574-98-7] YIKFFSNUOIJPSI-UHFFFAOYSA-N | $7.9\times10^{-7}$ $8.0\times10^{-7}$ $9.0\times10^{-7}$ | | Yaws (2003) Gharagheizi et al. (2012) Gharagheizi et al. (2010) | X Q Q | 237 246 |





Table A2.1: Alkanes (...continued)

| Substance Formula (Trivial Name) [CAS Registry Number] InChIKey | $H_s^{cp}$ (at $T^\ominus$) $\left[\dfrac{\text{mol}}{\text{m}^3\,\text{Pa}}\right]$ | $\dfrac{\text{d}\ln H_s^{cp}}{\text{d}(1/T)}$ [K] | Reference | Type | Note |
|---|---|---|---|---|---|
| 2,4,4,5,5-pentamethylheptane $C_{12}H_{26}$ [62199-73-9] TXGUYEKSMHZHSB-UHFFFAOYSA-N | $6.4\times10^{-7}$ $8.3\times10^{-7}$ $7.2\times10^{-7}$ | | Yaws (2003) Gharagheizi et al. (2012) Gharagheizi et al. (2010) | X Q Q | 237 246 |
| 3,3,4,4,5-pentamethylheptane $C_{12}H_{26}$ [62199-74-0] YFXNIDIHYOGTEN-UHFFFAOYSA-N | $6.1\times10^{-7}$ $1.1\times10^{-6}$ $7.2\times10^{-7}$ | | Yaws (2003) Gharagheizi et al. (2012) Gharagheizi et al. (2010) | X Q Q | 237 246 |
| 3,3,4,5,5-pentamethylheptane $C_{12}H_{26}$ [62199-75-1] NZIFKVSORPBQSX-UHFFFAOYSA-N | $6.1\times10^{-7}$ $1.2\times10^{-6}$ $7.2\times10^{-7}$ | | Yaws (2003) Gharagheizi et al. (2012) Gharagheizi et al. (2010) | X Q Q | 237 246 |
| 2,2,3-trimethyl-3-ethylheptane $C_{12}H_{26}$ [62199-04-6] CVXGCQXACVZXQR-UHFFFAOYSA-N | $6.9\times10^{-7}$ $9.3\times10^{-7}$ $7.5\times10^{-7}$ | | Yaws (2003) Gharagheizi et al. (2012) Gharagheizi et al. (2010) | X Q Q | 237 246 |
| 2,2,3-trimethyl-4-ethylheptane $C_{12}H_{26}$ [62199-05-7] COHRBFNOOUABCT-UHFFFAOYSA-N | $7.9\times10^{-7}$ $7.5\times10^{-7}$ $8.0\times10^{-7}$ | | Yaws (2003) Gharagheizi et al. (2012) Gharagheizi et al. (2010) | X Q Q | 237 246 |
| 2,2,3-trimethyl-5-ethylheptane $C_{12}H_{26}$ [62199-06-8] OOIGDPTZPWRGHN-UHFFFAOYSA-N | $8.0\times10^{-7}$ $7.3\times10^{-7}$ $8.0\times10^{-7}$ | | Yaws (2003) Gharagheizi et al. (2012) Gharagheizi et al. (2010) | X Q Q | 237 246 |
| 2,2,4-trimethyl-3-ethylheptane $C_{12}H_{26}$ [62199-07-9] TYGUUZVWCBYDKW-UHFFFAOYSA-N | $8.0\times10^{-7}$ $7.3\times10^{-7}$ $8.0\times10^{-7}$ | | Yaws (2003) Gharagheizi et al. (2012) Gharagheizi et al. (2010) | X Q Q | 237 246 |
| 2,2,4-trimethyl-4-ethylheptane $C_{12}H_{26}$ [62199-08-0] XMUXASPVOBQLPC-UHFFFAOYSA-N | $7.1\times10^{-7}$ $7.8\times10^{-7}$ $7.5\times10^{-7}$ | | Yaws (2003) Gharagheizi et al. (2012) Gharagheizi et al. (2010) | X Q Q | 237 246 |
| 2,2,4-trimethyl-5-ethylheptane $C_{12}H_{26}$ [62199-09-1] QADOHJKDWMLCQT-UHFFFAOYSA-N | $8.3\times10^{-7}$ $6.3\times10^{-7}$ $8.0\times10^{-7}$ | | Yaws (2003) Gharagheizi et al. (2012) Gharagheizi et al. (2010) | X Q Q | 237 246 |
| 2,2,5-trimethyl-3-ethylheptane $C_{12}H_{26}$ [62199-10-4] SUDWTPMZYLBGKW-UHFFFAOYSA-N | $8.1\times10^{-7}$ $6.9\times10^{-7}$ $8.0\times10^{-7}$ | | Yaws (2003) Gharagheizi et al. (2012) Gharagheizi et al. (2010) | X Q Q | 237 246 |



Table A2.1: Alkanes (...continued)

| Substance<br>Formula<br>(Trivial Name)<br>[CAS Registry Number]<br>InChIKey | $H_s^{cp}$<br>(at $T^\ominus$)<br>$\left[\dfrac{\mathrm{mol}}{\mathrm{m^3\,Pa}}\right]$ | $\dfrac{\mathrm{d}\ln H_s^{cp}}{\mathrm{d}(1/T)}$<br><br>[K] | Reference | Type | Note |
|---|---|---|---|---|---|
| 2,2,5-trimethyl-4-ethylheptane<br>$C_{12}H_{26}$<br>[62199-11-5]<br>GDPLKYYKURLRKJ-UHFFFAOYSA-N | $8.4\times10^{-7}$<br>$6.2\times10^{-7}$<br>$8.0\times10^{-7}$ | | Yaws (2003)<br>Gharagheizi et al. (2012)<br>Gharagheizi et al. (2010) | X<br>Q<br>Q | 237<br><br>246 |
| 2,2,5-trimethyl-5-ethylheptane<br>$C_{12}H_{26}$<br>[62199-12-6]<br>KJWMMFYTYSUWMH-UHFFFAOYSA-N | $7.4\times10^{-7}$<br>$6.7\times10^{-7}$<br>$7.5\times10^{-7}$ | | Yaws (2003)<br>Gharagheizi et al. (2012)<br>Gharagheizi et al. (2010) | X<br>Q<br>Q | 237<br><br>246 |
| 2,2,6-trimethyl-3-ethylheptane<br>$C_{12}H_{26}$<br>[62199-13-7]<br>RIIDJOQRWGTQEN-UHFFFAOYSA-N | $8.1\times10^{-7}$<br>$6.9\times10^{-7}$<br>$8.0\times10^{-7}$ | | Yaws (2003)<br>Gharagheizi et al. (2012)<br>Gharagheizi et al. (2010) | X<br>Q<br>Q | 237<br><br>246 |
| 2,2,6-trimethyl-4-ethylheptane<br>$C_{12}H_{26}$<br>[62199-14-8]<br>QGYSVVWKEDKPTP-UHFFFAOYSA-N | $8.9\times10^{-7}$<br>$5.2\times10^{-7}$<br>$8.0\times10^{-7}$ | | Yaws (2003)<br>Gharagheizi et al. (2012)<br>Gharagheizi et al. (2010) | X<br>Q<br>Q | 237<br><br>246 |
| 2,2,6-trimethyl-5-ethylheptane<br>$C_{12}H_{26}$<br>[62199-15-9]<br>LYZRWXZZVRWJHX-UHFFFAOYSA-N | $8.3\times10^{-7}$<br>$6.3\times10^{-7}$<br>$8.0\times10^{-7}$ | | Yaws (2003)<br>Gharagheizi et al. (2012)<br>Gharagheizi et al. (2010) | X<br>Q<br>Q | 237<br><br>246 |
| 2,3,3-trimethyl-4-ethylheptane<br>$C_{12}H_{26}$<br>[62199-16-0]<br>VIIXOXRHJWHIBP-UHFFFAOYSA-N | $7.6\times10^{-7}$<br>$9.0\times10^{-7}$<br>$8.0\times10^{-7}$ | | Yaws (2003)<br>Gharagheizi et al. (2012)<br>Gharagheizi et al. (2010) | X<br>Q<br>Q | 237<br><br>246 |
| 2,3,3-trimethyl-5-ethylheptane<br>$C_{12}H_{26}$<br>[62199-17-1]<br>JHGOLBUJAIJLCX-UHFFFAOYSA-N | $7.9\times10^{-7}$<br>$7.8\times10^{-7}$<br>$8.0\times10^{-7}$ | | Yaws (2003)<br>Gharagheizi et al. (2012)<br>Gharagheizi et al. (2010) | X<br>Q<br>Q | 237<br><br>246 |
| 2,3,4-trimethyl-3-ethylheptane<br>$C_{12}H_{26}$<br>[62199-18-2]<br>YZXMDESZKHUODO-UHFFFAOYSA-N | $7.5\times10^{-7}$<br>$1.0\times10^{-6}$<br>$8.0\times10^{-7}$ | | Yaws (2003)<br>Gharagheizi et al. (2012)<br>Gharagheizi et al. (2010) | X<br>Q<br>Q | 237<br><br>246 |
| 2,3,4-trimethyl-4-ethylheptane<br>$C_{12}H_{26}$<br>[62198-55-4]<br>QDWZIZQKVLESDK-UHFFFAOYSA-N | $7.7\times10^{-7}$<br>$8.7\times10^{-7}$<br>$8.0\times10^{-7}$ | | Yaws (2003)<br>Gharagheizi et al. (2012)<br>Gharagheizi et al. (2010) | X<br>Q<br>Q | 237<br><br>246 |
| 2,3,4-trimethyl-5-ethylheptane<br>$C_{12}H_{26}$<br>[62198-56-5]<br>URFFXJPJMCMXFR-UHFFFAOYSA-N | $8.5\times10^{-7}$<br>$8.4\times10^{-7}$<br>$8.9\times10^{-7}$ | | Yaws (2003)<br>Gharagheizi et al. (2012)<br>Gharagheizi et al. (2010) | X<br>Q<br>Q | 237<br><br>246 |





Table A2.1: Alkanes (...continued)

| Substance Formula (Trivial Name) [CAS Registry Number] InChIKey | $H_s^{cp}$ (at $T^\ominus$) $\left[\dfrac{\text{mol}}{\text{m}^3\,\text{Pa}}\right]$ | $\dfrac{\mathrm{d}\ln H_s^{cp}}{\mathrm{d}(1/T)}$ [K] | Reference | Type | Note |
|---|---|---|---|---|---|
| 2,3,5-trimethyl-3-ethylheptane $C_{12}H_{26}$ [62198-57-6] QHACFORXSGHTJG-UHFFFAOYSA-N | $7.7\times10^{-7}$ $8.6\times10^{-7}$ $8.0\times10^{-7}$ | | Yaws (2003) Gharagheizi et al. (2012) Gharagheizi et al. (2010) | X Q Q | 237 246 |
| 2,3,5-trimethyl-4-ethylheptane $C_{12}H_{26}$ [62198-58-7] UEZAASCWBAVHRO-UHFFFAOYSA-N | $8.6\times10^{-7}$ $8.2\times10^{-7}$ $8.9\times10^{-7}$ | | Yaws (2003) Gharagheizi et al. (2012) Gharagheizi et al. (2010) | X Q Q | 237 246 |
| 2,3,5-trimethyl-5-ethylheptane $C_{12}H_{26}$ [62198-59-8] LTSLSBVPXXAMBM-UHFFFAOYSA-N | $7.9\times10^{-7}$ $7.8\times10^{-7}$ $8.0\times10^{-7}$ | | Yaws (2003) Gharagheizi et al. (2012) Gharagheizi et al. (2010) | X Q Q | 237 246 |
| 2,3,6-trimethyl-3-ethylheptane $C_{12}H_{26}$ [62198-60-1] KOVQAKVRXODFNL-UHFFFAOYSA-N | $7.8\times10^{-7}$ $8.2\times10^{-7}$ $8.0\times10^{-7}$ | | Yaws (2003) Gharagheizi et al. (2012) Gharagheizi et al. (2010) | X Q Q | 237 246 |
| 2,3,6-trimethyl-4-ethylheptane $C_{12}H_{26}$ [62198-61-2] RBVKVRIWSQSXPO-UHFFFAOYSA-N | $8.9\times10^{-7}$ $6.9\times10^{-7}$ $8.9\times10^{-7}$ | | Yaws (2003) Gharagheizi et al. (2012) Gharagheizi et al. (2010) | X Q Q | 237 246 |
| 2,3,6-trimethyl-5-ethylheptane $C_{12}H_{26}$ [62198-62-3] JXCQTSFHNKTOLJ-UHFFFAOYSA-N | $8.7\times10^{-7}$ $7.5\times10^{-7}$ $8.9\times10^{-7}$ | | Yaws (2003) Gharagheizi et al. (2012) Gharagheizi et al. (2010) | X Q Q | 237 246 |
| 2,4,4-trimethyl-3-ethylheptane $C_{12}H_{26}$ [62198-63-4] BNFUBMCCIACHFX-UHFFFAOYSA-N | $7.9\times10^{-7}$ $7.8\times10^{-7}$ $8.0\times10^{-7}$ | | Yaws (2003) Gharagheizi et al. (2012) Gharagheizi et al. (2010) | X Q Q | 237 246 |
| 2,4,4-trimethyl-5-ethylheptane $C_{12}H_{26}$ [62198-64-5] YSCJXRHNKUXHLH-UHFFFAOYSA-N | $8.0\times10^{-7}$ $7.3\times10^{-7}$ $8.0\times10^{-7}$ | | Yaws (2003) Gharagheizi et al. (2012) Gharagheizi et al. (2010) | X Q Q | 237 246 |
| 2,4,5-trimethyl-3-ethylheptane $C_{12}H_{26}$ [62198-65-6] JFKSTTDMBQAPCT-UHFFFAOYSA-N | $8.6\times10^{-7}$ $8.2\times10^{-7}$ $8.9\times10^{-7}$ | | Yaws (2003) Gharagheizi et al. (2012) Gharagheizi et al. (2010) | X Q Q | 237 246 |
| 2,4,5-trimethyl-4-ethylheptane $C_{12}H_{26}$ [62198-66-7] HSVLOEWHJNAWLC-UHFFFAOYSA-N | $7.8\times10^{-7}$ $8.0\times10^{-7}$ $8.0\times10^{-7}$ | | Yaws (2003) Gharagheizi et al. (2012) Gharagheizi et al. (2010) | X Q Q | 237 246 |





Table A2.1: Alkanes (. . . continued)

| Substance<br>Formula<br>(Trivial Name)<br>[CAS Registry Number]<br>InChIKey | $H_s^{cp}$<br>(at $T^\ominus$)<br>$\left[\dfrac{\mathrm{mol}}{\mathrm{m}^3\,\mathrm{Pa}}\right]$ | $\dfrac{\mathrm{d}\ln H_s^{cp}}{\mathrm{d}(1/T)}$<br><br>[K] | Reference | Type | Note |
|---|---|---|---|---|---|
| 2,4,5-trimethyl-5-ethylheptane<br>$C_{12}H_{26}$<br>[62198-67-8]<br>JAUBFUIZENVVAG-UHFFFAOYSA-N | $7.7\times10^{-7}$<br>$8.5\times10^{-7}$<br>$8.0\times10^{-7}$ | | Yaws (2003)<br>Gharagheizi et al. (2012)<br>Gharagheizi et al. (2010) | X<br>Q<br>Q | 237<br><br>246 |
| 2,4,6-trimethyl-3-ethylheptane<br>$C_{12}H_{26}$<br>[62198-68-9]<br>KQBJYFJWQOUHCI-UHFFFAOYSA-N | $8.8\times10^{-7}$<br>$7.1\times10^{-7}$<br>$8.9\times10^{-7}$ | | Yaws (2003)<br>Gharagheizi et al. (2012)<br>Gharagheizi et al. (2010) | X<br>Q<br>Q | 237<br><br>246 |
| 2,4,6-trimethyl-4-ethylheptane<br>$C_{12}H_{26}$<br>[62198-69-0]<br>LPNPQCHEMXFBSW-UHFFFAOYSA-N | $8.4\times10^{-7}$<br>$6.0\times10^{-7}$<br>$8.0\times10^{-7}$ | | Yaws (2003)<br>Gharagheizi et al. (2012)<br>Gharagheizi et al. (2010) | X<br>Q<br>Q | 237<br><br>246 |
| 2,5,5-trimethyl-3-ethylheptane<br>$C_{12}H_{26}$<br>[62198-70-3]<br>MARFNXCIPCRAJU-UHFFFAOYSA-N | $8.1\times10^{-7}$<br>$6.9\times10^{-7}$<br>$8.0\times10^{-7}$ | | Yaws (2003)<br>Gharagheizi et al. (2012)<br>Gharagheizi et al. (2010) | X<br>Q<br>Q | 237<br><br>246 |
| 2,5,5-trimethyl-4-ethylheptane<br>$C_{12}H_{26}$<br>[62198-71-4]<br>HGYOEMXZYDBOKQ-UHFFFAOYSA-N | $8.0\times10^{-7}$<br>$7.3\times10^{-7}$<br>$8.0\times10^{-7}$ | | Yaws (2003)<br>Gharagheizi et al. (2012)<br>Gharagheizi et al. (2010) | X<br>Q<br>Q | 237<br><br>246 |
| 3,3,4-trimethyl-4-ethylheptane<br>$C_{12}H_{26}$<br>[62198-72-5]<br>QLOXTBGCIUOWIG-UHFFFAOYSA-N | $6.8\times10^{-7}$<br>$1.1\times10^{-6}$<br>$7.5\times10^{-7}$ | | Yaws (2003)<br>Gharagheizi et al. (2012)<br>Gharagheizi et al. (2010) | X<br>Q<br>Q | 237<br><br>246 |
| 3,3,4-trimethyl-5-ethylheptane<br>$C_{12}H_{26}$<br>[62198-73-6]<br>CUWOOQVXUDOHOC-UHFFFAOYSA-N | $7.6\times10^{-7}$<br>$9.0\times10^{-7}$<br>$8.0\times10^{-7}$ | | Yaws (2003)<br>Gharagheizi et al. (2012)<br>Gharagheizi et al. (2010) | X<br>Q<br>Q | 237<br><br>246 |
| 3,3,5-trimethyl-4-ethylheptane<br>$C_{12}H_{26}$<br>[62198-74-7]<br>QXCHZKHHIUTBOU-UHFFFAOYSA-N | $7.7\times10^{-7}$<br>$8.5\times10^{-7}$<br>$8.0\times10^{-7}$ | | Yaws (2003)<br>Gharagheizi et al. (2012)<br>Gharagheizi et al. (2010) | X<br>Q<br>Q | 237<br><br>246 |
| 3,3,5-trimethyl-5-ethylheptane<br>$C_{12}H_{26}$<br>[62198-75-8]<br>GCWLDOMARVHVOZ-UHFFFAOYSA-N | $6.8\times10^{-7}$<br>$9.8\times10^{-7}$<br>$7.5\times10^{-7}$ | | Yaws (2003)<br>Gharagheizi et al. (2012)<br>Gharagheizi et al. (2010) | X<br>Q<br>Q | 237<br><br>246 |
| 3,4,4-trimethyl-3-ethylheptane<br>$C_{12}H_{26}$<br>[62198-76-9]<br>SIIFYRMARKBLER-UHFFFAOYSA-N | $6.8\times10^{-7}$<br>$1.1\times10^{-6}$<br>$7.5\times10^{-7}$ | | Yaws (2003)<br>Gharagheizi et al. (2012)<br>Gharagheizi et al. (2010) | X<br>Q<br>Q | 237<br><br>246 |



Table A2.1: Alkanes (...continued)

| Substance Formula (Trivial Name) [CAS Registry Number] InChIKey | $H_s^{cp}$ (at $T^\ominus$) $\left[ \dfrac{\text{mol}}{\text{m}^3\,\text{Pa}} \right]$ | $\dfrac{\text{d} \ln H_s^{cp}}{\text{d}(1/T)}$ [K] | Reference | Type | Note |
|---|---|---|---|---|---|
| 3,4,4-trimethyl-5-ethylheptane $C_{12}H_{26}$ [62198-77-0] XSLWCLUAFIVTLD-UHFFFAOYSA-N | $7.6 \times 10^{-7}$ $9.8 \times 10^{-7}$ $8.0 \times 10^{-7}$ | | Yaws (2003) Gharagheizi et al. (2012) Gharagheizi et al. (2010) | X Q Q | 237 246 |
| 3,4,5-trimethyl-3-ethylheptane $C_{12}H_{26}$ [62198-78-1] ZHILUIHUWKDTTM-UHFFFAOYSA-N | $7.6 \times 10^{-7}$ $9.8 \times 10^{-7}$ $8.0 \times 10^{-7}$ | | Yaws (2003) Gharagheizi et al. (2012) Gharagheizi et al. (2010) | X Q Q | 237 246 |
| 3,4,5-trimethyl-4-ethylheptane $C_{12}H_{26}$ [62198-79-2] SZOKDASFURUJDF-UHFFFAOYSA-N | $7.5 \times 10^{-7}$ $1.1 \times 10^{-6}$ $8.0 \times 10^{-7}$ | | Yaws (2003) Gharagheizi et al. (2012) Gharagheizi et al. (2010) | X Q Q | 237 246 |
| 2,2-dimethyl-4-propylheptane $C_{12}H_{26}$ DWEXNJRCPKKJBQ-UHFFFAOYSA-N | $9.2 \times 10^{-7}$ $5.8 \times 10^{-7}$ $8.7 \times 10^{-7}$ | | Yaws (2003) Gharagheizi et al. (2012) Gharagheizi et al. (2010) | X Q Q | 237 246 |
| 2,3-dimethyl-4-propylheptane $C_{12}H_{26}$ [62185-30-2] AZUDVOIXPNEOLR-UHFFFAOYSA-N | $9.5 \times 10^{-7}$ $7.7 \times 10^{-7}$ $9.2 \times 10^{-7}$ | | Yaws (2003) Gharagheizi et al. (2012) Gharagheizi et al. (2010) | X Q Q | 237 246 |
| 2,4-dimethyl-4-propylheptane $C_{12}H_{26}$ [62185-31-3] FDVJFPZZPSPQAO-UHFFFAOYSA-N | $8.9 \times 10^{-7}$ $6.7 \times 10^{-7}$ $8.7 \times 10^{-7}$ | | Yaws (2003) Gharagheizi et al. (2012) Gharagheizi et al. (2010) | X Q Q | 237 246 |
| 2,5-dimethyl-4-propylheptane $C_{12}H_{26}$ [62185-32-4] PJXVVWKRCASATD-UHFFFAOYSA-N | $9.7 \times 10^{-7}$ $6.8 \times 10^{-7}$ $9.2 \times 10^{-7}$ | | Yaws (2003) Gharagheizi et al. (2012) Gharagheizi et al. (2010) | X Q Q | 237 246 |
| 2,6-dimethyl-4-propylheptane $C_{12}H_{26}$ [62185-33-5] UFJZYHSURMZROL-UHFFFAOYSA-N | $1.0 \times 10^{-6}$ $5.9 \times 10^{-7}$ $9.2 \times 10^{-7}$ | | Yaws (2003) Gharagheizi et al. (2012) Gharagheizi et al. (2010) | X Q Q | 237 246 |
| 3,3-dimethyl-4-propylheptane $C_{12}H_{26}$ [62185-34-6] XVWBLUKOUJTCDO-UHFFFAOYSA-N | $8.5 \times 10^{-7}$ $8.2 \times 10^{-7}$ $8.7 \times 10^{-7}$ | | Yaws (2003) Gharagheizi et al. (2012) Gharagheizi et al. (2010) | X Q Q | 237 246 |
| 3,4-dimethyl-4-propylheptane $C_{12}H_{26}$ [62185-35-7] UGSAJWKYCFFZRD-UHFFFAOYSA-N | $8.4 \times 10^{-7}$ $8.6 \times 10^{-7}$ $8.7 \times 10^{-7}$ | | Yaws (2003) Gharagheizi et al. (2012) Gharagheizi et al. (2010) | X Q Q | 237 246 |



Table A2.1: Alkanes (...continued)

| Substance Formula (Trivial Name) [CAS Registry Number] InChIKey | $H_s^{cp}$ (at $T^{\ominus}$) $\left[\dfrac{\mathrm{mol}}{\mathrm{m^3\,Pa}}\right]$ | $\dfrac{\mathrm{d}\ln H_s^{cp}}{\mathrm{d}(1/T)}$ [K] | Reference | Type | Note |
|---|---|---|---|---|---|
| 3,5-dimethyl-4-propylheptane $C_{12}H_{26}$ [62185-36-8] GKVRDSIZILZQJE-UHFFFAOYSA-N | $9.4\times10^{-7}$ $8.1\times10^{-7}$ $9.2\times10^{-7}$ | | Yaws (2003) Gharagheizi et al. (2012) Gharagheizi et al. (2010) | X Q Q | 237 246 |
| 2,2-dimethyl-3-isopropylheptane $C_{12}H_{26}$ [62185-37-9] XPOSVKCTMHOWTN-UHFFFAOYSA-N | $8.1\times10^{-7}$ $6.9\times10^{-7}$ $8.0\times10^{-7}$ | | Yaws (2003) Gharagheizi et al. (2012) Gharagheizi et al. (2010) | X Q Q | 237 246 |
| 2,2-dimethyl-4-isopropylheptane $C_{12}H_{26}$ [62185-38-0] HFRVMJCEWUXCMP-UHFFFAOYSA-N | $8.7\times10^{-7}$ $5.5\times10^{-7}$ $8.0\times10^{-7}$ | | Yaws (2003) Gharagheizi et al. (2012) Gharagheizi et al. (2010) | X Q Q | 237 246 |
| 2,3-dimethyl-3-isopropylheptane $C_{12}H_{26}$ [62185-39-1] IAWDNHPMOJTEEN-UHFFFAOYSA-N | $7.6\times10^{-7}$ $9.5\times10^{-7}$ $8.0\times10^{-7}$ | | Yaws (2003) Gharagheizi et al. (2012) Gharagheizi et al. (2010) | X Q Q | 237 246 |
| 2,3-dimethyl-4-isopropylheptane $C_{12}H_{26}$ [62185-40-4] GEQDAUMUCVOARW-UHFFFAOYSA-N | $8.8\times10^{-7}$ $7.3\times10^{-7}$ $8.9\times10^{-7}$ | | Yaws (2003) Gharagheizi et al. (2012) Gharagheizi et al. (2010) | X Q Q | 237 246 |
| 2,4-dimethyl-3-isopropylheptane $C_{12}H_{26}$ [62185-41-5] MWVRZCOXSIAHLW-UHFFFAOYSA-N | $8.8\times10^{-7}$ $7.3\times10^{-7}$ $8.9\times10^{-7}$ | | Yaws (2003) Gharagheizi et al. (2012) Gharagheizi et al. (2010) | X Q Q | 237 246 |
| 2,4-dimethyl-4-isopropylheptane $C_{12}H_{26}$ [62185-42-6] MTPOSRQPOBWFQC-UHFFFAOYSA-N | $8.0\times10^{-7}$ $7.3\times10^{-7}$ $8.0\times10^{-7}$ | | Yaws (2003) Gharagheizi et al. (2012) Gharagheizi et al. (2010) | X Q Q | 237 246 |
| 2,5-dimethyl-3-isopropylheptane $C_{12}H_{26}$ [62185-43-7] XSKMFUBSJXETFW-UHFFFAOYSA-N | $8.9\times10^{-7}$ $6.9\times10^{-7}$ $8.9\times10^{-7}$ | | Yaws (2003) Gharagheizi et al. (2012) Gharagheizi et al. (2010) | X Q Q | 237 246 |
| 2,5-dimethyl-4-isopropylheptane $C_{12}H_{26}$ [62185-44-8] OOYWQZAQFKOCKA-UHFFFAOYSA-N | $9.0\times10^{-7}$ $6.7\times10^{-7}$ $8.9\times10^{-7}$ | | Yaws (2003) Gharagheizi et al. (2012) Gharagheizi et al. (2010) | X Q Q | 237 246 |
| 2,6-dimethyl-3-isopropylheptane $C_{12}H_{26}$ [62185-45-9] NOAGKIODSFDOTA-UHFFFAOYSA-N | $8.8\times10^{-7}$ $7.1\times10^{-7}$ $8.9\times10^{-7}$ | | Yaws (2003) Gharagheizi et al. (2012) Gharagheizi et al. (2010) | X Q Q | 237 246 |





Table A2.1: Alkanes (...continued)

| Substance<br>Formula<br>(Trivial Name)<br>[CAS Registry Number]<br>InChIKey | $H_s^{cp}$<br>(at $T^\ominus$)<br>$\left[\dfrac{\mathrm{mol}}{\mathrm{m^3\,Pa}}\right]$ | $\dfrac{\mathrm{d}\ln H_s^{cp}}{\mathrm{d}(1/T)}$<br><br>[K] | Reference | Type | Note |
|---|---|---|---|---|---|
| 2,6-dimethyl-4-isopropylheptane<br>$C_{12}H_{26}$<br>[35866-89-8]<br>BIBHNAJLMRUJLS-UHFFFAOYSA-N | $9.4\times10^{-7}$<br>$5.6\times10^{-7}$<br>$8.9\times10^{-7}$ | | Yaws (2003)<br>Gharagheizi et al. (2012)<br>Gharagheizi et al. (2010) | X<br>Q<br>Q | 237<br><br>246 |
| 3,3-dimethyl-4-isopropylheptane<br>$C_{12}H_{26}$<br>[62185-46-0]<br>MACIOLXBYDNTTI-UHFFFAOYSA-N | $7.9\times10^{-7}$<br>$7.8\times10^{-7}$<br>$8.0\times10^{-7}$ | | Yaws (2003)<br>Gharagheizi et al. (2012)<br>Gharagheizi et al. (2010) | X<br>Q<br>Q | 237<br><br>246 |
| 3,4-dimethyl-4-isopropylheptane<br>$C_{12}H_{26}$<br>[62185-47-1]<br>QFKXRROMVRXZIF-UHFFFAOYSA-N | $7.6\times10^{-7}$<br>$9.5\times10^{-7}$<br>$8.0\times10^{-7}$ | | Yaws (2003)<br>Gharagheizi et al. (2012)<br>Gharagheizi et al. (2010) | X<br>Q<br>Q | 237<br><br>246 |
| 3,5-dimethyl-4-isopropylheptane<br>$C_{12}H_{26}$<br>[62198-89-4]<br>XJRRTENBDSNHQB-UHFFFAOYSA-N | $8.7\times10^{-7}$<br>$7.7\times10^{-7}$<br>$8.9\times10^{-7}$ | | Yaws (2003)<br>Gharagheizi et al. (2012)<br>Gharagheizi et al. (2010) | X<br>Q<br>Q | 237<br><br>246 |
| 2-methyl-3,3-diethylheptane<br>$C_{12}H_{26}$<br>[62198-90-7]<br>VBKOVPDJLQIMSZ-UHFFFAOYSA-N | $8.3\times10^{-7}$<br>$1.0\times10^{-6}$<br>$8.7\times10^{-7}$ | | Yaws (2003)<br>Gharagheizi et al. (2012)<br>Gharagheizi et al. (2010) | X<br>Q<br>Q | 237<br><br>246 |
| 2-methyl-3,4-diethylheptane<br>$C_{12}H_{26}$<br>[62198-91-8]<br>PRSHREOBBOOUSO-UHFFFAOYSA-N | $9.4\times10^{-7}$<br>$7.9\times10^{-7}$<br>$9.2\times10^{-7}$ | | Yaws (2003)<br>Gharagheizi et al. (2012)<br>Gharagheizi et al. (2010) | X<br>Q<br>Q | 237<br><br>246 |
| 2-methyl-3,5-diethylheptane<br>$C_{12}H_{26}$<br>[62198-92-9]<br>KZTGLPPJUZINLQ-UHFFFAOYSA-N | $9.4\times10^{-7}$<br>$7.9\times10^{-7}$<br>$9.2\times10^{-7}$ | | Yaws (2003)<br>Gharagheizi et al. (2012)<br>Gharagheizi et al. (2010) | X<br>Q<br>Q | 237<br><br>246 |
| 2-methyl-4,4-diethylheptane<br>$C_{12}H_{26}$<br>[62198-93-0]<br>VJTVFYMTOXDIID-UHFFFAOYSA-N | $8.6\times10^{-7}$<br>$7.7\times10^{-7}$<br>$8.7\times10^{-7}$ | | Yaws (2003)<br>Gharagheizi et al. (2012)<br>Gharagheizi et al. (2010) | X<br>Q<br>Q | 237<br><br>246 |
| 2-methyl-4,5-diethylheptane<br>$C_{12}H_{26}$<br>[62198-94-1]<br>SRLQJPCYGKMUSM-UHFFFAOYSA-N | $9.5\times10^{-7}$<br>$7.5\times10^{-7}$<br>$9.2\times10^{-7}$ | | Yaws (2003)<br>Gharagheizi et al. (2012)<br>Gharagheizi et al. (2010) | X<br>Q<br>Q | 237<br><br>246 |
| 2-methyl-5,5-diethylheptane<br>$C_{12}H_{26}$<br>[62198-95-2]<br>XPAWKGLXSPHNAN-UHFFFAOYSA-N | $8.4\times10^{-7}$<br>$8.6\times10^{-7}$<br>$8.7\times10^{-7}$ | | Yaws (2003)<br>Gharagheizi et al. (2012)<br>Gharagheizi et al. (2010) | X<br>Q<br>Q | 237<br><br>246 |



Table A2.1: Alkanes (...continued)

| Substance<br>Formula<br>(Trivial Name)<br>[CAS Registry Number]<br>InChIKey | $H_s^{cp}$<br>(at $T^{\ominus}$)<br>$\left[\dfrac{\text{mol}}{\text{m}^3\,\text{Pa}}\right]$ | $\dfrac{\text{d}\ln H_s^{cp}}{\text{d}(1/T)}$<br><br>[K] | Reference | Type | Note |
|---|---|---|---|---|---|
| 3-methyl-3,4-diethylheptane<br>$C_{12}H_{26}$<br>[62198-96-3]<br>VRVVWJIPKQWWJB-UHFFFAOYSA-N | $8.3\times10^{-7}$<br>$9.4\times10^{-7}$<br>$8.7\times10^{-7}$ | | Yaws (2003)<br>Gharagheizi et al. (2012)<br>Gharagheizi et al. (2010) | X<br>Q<br>Q | 237<br><br>246 |
| 3-methyl-3,5-diethylheptane<br>$C_{12}H_{26}$<br>[62198-97-4]<br>RGRSPNWZJFXFAN-UHFFFAOYSA-N | $8.5\times10^{-7}$<br>$8.4\times10^{-7}$<br>$8.7\times10^{-7}$ | | Yaws (2003)<br>Gharagheizi et al. (2012)<br>Gharagheizi et al. (2010) | X<br>Q<br>Q | 237<br><br>246 |
| 3-methyl-4,4-diethylheptane<br>$C_{12}H_{26}$<br>[62198-98-5]<br>WSBKWCXGZHVUHW-UHFFFAOYSA-N | $8.3\times10^{-7}$<br>$1.0\times10^{-6}$<br>$8.7\times10^{-7}$ | | Yaws (2003)<br>Gharagheizi et al. (2012)<br>Gharagheizi et al. (2010) | X<br>Q<br>Q | 237<br><br>246 |
| 3-methyl-4,5-diethylheptane<br>$C_{12}H_{26}$<br>[62198-99-6]<br>NVBVLQDXCRDBBU-UHFFFAOYSA-N | $9.3\times10^{-7}$<br>$8.6\times10^{-7}$<br>$9.2\times10^{-7}$ | | Yaws (2003)<br>Gharagheizi et al. (2012)<br>Gharagheizi et al. (2010) | X<br>Q<br>Q | 237<br><br>246 |
| 3-methyl-5,5-diethylheptane<br>$C_{12}H_{26}$<br>[62199-00-2]<br>JOJSCBROPYJHJX-UHFFFAOYSA-N | $8.4\times10^{-7}$<br>$8.9\times10^{-7}$<br>$8.7\times10^{-7}$ | | Yaws (2003)<br>Gharagheizi et al. (2012)<br>Gharagheizi et al. (2010) | X<br>Q<br>Q | 237<br><br>246 |
| 4-methyl-3,3-diethylheptane<br>$C_{12}H_{26}$<br>[62199-01-3]<br>KGSWPEXUAXYDMO-UHFFFAOYSA-N | $8.2\times10^{-7}$<br>$1.1\times10^{-6}$<br>$8.7\times10^{-7}$ | | Yaws (2003)<br>Gharagheizi et al. (2012)<br>Gharagheizi et al. (2010) | X<br>Q<br>Q | 237<br><br>246 |
| 4-methyl-3,4-diethylheptane<br>$C_{12}H_{26}$<br>[62199-02-4]<br>RAFABRWSQNCWEY-UHFFFAOYSA-N | $8.3\times10^{-7}$<br>$9.4\times10^{-7}$<br>$8.7\times10^{-7}$ | | Yaws (2003)<br>Gharagheizi et al. (2012)<br>Gharagheizi et al. (2010) | X<br>Q<br>Q | 237<br><br>246 |
| 4-methyl-3,5-diethylheptane<br>$C_{12}H_{26}$<br>[62199-03-5]<br>XUVBJZPYCGZQDK-UHFFFAOYSA-N | $9.2\times10^{-7}$<br>$9.1\times10^{-7}$<br>$9.2\times10^{-7}$ | | Yaws (2003)<br>Gharagheizi et al. (2012)<br>Gharagheizi et al. (2010) | X<br>Q<br>Q | 237<br><br>246 |
| 2-methyl-4-*tert*-butylheptane<br>$C_{12}H_{26}$<br>[62185-23-3]<br>HIHQJZLWJQHTEJ-UHFFFAOYSA-N | $8.4\times10^{-7}$<br>$6.0\times10^{-7}$<br>$8.0\times10^{-7}$ | | Yaws (2003)<br>Gharagheizi et al. (2012)<br>Gharagheizi et al. (2010) | X<br>Q<br>Q | 237<br><br>246 |
| 3-methyl-4-*tert*-butylheptane<br>$C_{12}H_{26}$<br>[62185-24-4]<br>IKKVFRPBQQTTKP-UHFFFAOYSA-N | $8.1\times10^{-7}$<br>$6.9\times10^{-7}$<br>$8.0\times10^{-7}$ | | Yaws (2003)<br>Gharagheizi et al. (2012)<br>Gharagheizi et al. (2010) | X<br>Q<br>Q | 237<br><br>246 |





Table A2.1: Alkanes (. . . continued)

| Substance Formula (Trivial Name) [CAS Registry Number] InChIKey | $H_s^{cp}$ (at $T^\ominus$) $\left[\dfrac{\text{mol}}{\text{m}^3\,\text{Pa}}\right]$ | $\dfrac{\text{d}\ln H_s^{cp}}{\text{d}(1/T)}$ [K] | Reference | Type | Note |
|---|---|---|---|---|---|
| 4-methyl-4-*tert*-butylheptane<br>$C_{12}H_{26}$<br>[62185-25-5]<br>JABLMQZBFVBNBS-UHFFFAOYSA-N | $7.0\times10^{-7}$<br>$8.7\times10^{-7}$<br>$7.5\times10^{-7}$ | | Yaws (2003)<br>Gharagheizi et al. (2012)<br>Gharagheizi et al. (2010) | X<br>Q<br>Q | 237<br><br>246 |
| 3-ethyl-4-propylheptane<br>$C_{12}H_{26}$<br>[62185-26-6]<br>FPODIYZXVCXXQE-UHFFFAOYSA-N | $1.0\times10^{-6}$<br>$8.1\times10^{-7}$<br>$1.0\times10^{-6}$ | | Yaws (2003)<br>Gharagheizi et al. (2012)<br>Gharagheizi et al. (2010) | X<br>Q<br>Q | 237<br><br>246 |
| 4-ethyl-4-propylheptane<br>$C_{12}H_{26}$<br>[17312-43-5]<br>JTZJHTZCYBYCSZ-UHFFFAOYSA-N | $9.3\times10^{-7}$<br>$8.3\times10^{-7}$<br>$1.0\times10^{-6}$ | | Yaws (2003)<br>Gharagheizi et al. (2012)<br>Gharagheizi et al. (2010) | X<br>Q<br>Q | 237<br><br>246 |
| 3-ethyl-4-isopropylheptane<br>$C_{12}H_{26}$<br>[62185-27-7]<br>VENQCMDSFBOTMS-UHFFFAOYSA-N | $9.4\times10^{-7}$<br>$7.9\times10^{-7}$<br>$9.2\times10^{-7}$ | | Yaws (2003)<br>Gharagheizi et al. (2012)<br>Gharagheizi et al. (2010) | X<br>Q<br>Q | 237<br><br>246 |
| 4-ethyl-4-isopropylheptane<br>$C_{12}H_{26}$<br>[62185-28-8]<br>IKRHMKIYGHBTGZ-UHFFFAOYSA-N | $8.3\times10^{-7}$<br>$9.2\times10^{-7}$<br>$8.7\times10^{-7}$ | | Yaws (2003)<br>Gharagheizi et al. (2012)<br>Gharagheizi et al. (2010) | X<br>Q<br>Q | 237<br><br>246 |
| 2,2,3,3,4,4-hexamethylhexane<br>$C_{12}H_{26}$<br>[62185-11-9]<br>ACZDZCRQHVWGEF-UHFFFAOYSA-N | $4.9\times10^{-7}$<br>$1.5\times10^{-6}$<br>$6.7\times10^{-7}$ | | Yaws (2003)<br>Gharagheizi et al. (2012)<br>Gharagheizi et al. (2010) | X<br>Q<br>Q | 237<br><br>246 |
| 2,2,3,3,4,5-hexamethylhexane<br>$C_{12}H_{26}$<br>[62185-12-0]<br>IFHVEHXJQMYGHH-UHFFFAOYSA-N | $5.8\times10^{-7}$<br>$9.1\times10^{-7}$<br>$7.4\times10^{-7}$ | | Yaws (2003)<br>Gharagheizi et al. (2012)<br>Gharagheizi et al. (2010) | X<br>Q<br>Q | 237<br><br>246 |
| 2,2,3,3,5,5-hexamethylhexane<br>$C_{12}H_{26}$<br>[60302-24-1]<br>YFGLJPQISFJQSO-UHFFFAOYSA-N | $5.4\times10^{-7}$<br>$8.2\times10^{-7}$<br>$6.7\times10^{-7}$ | | Yaws (2003)<br>Gharagheizi et al. (2012)<br>Gharagheizi et al. (2010) | X<br>Q<br>Q | 237<br><br>246 |
| 2,2,3,4,4,5-hexamethylhexane<br>$C_{12}H_{26}$<br>[62185-13-1]<br>QJVUKMHIEDVCDT-UHFFFAOYSA-N | $5.6\times10^{-7}$<br>$1.1\times10^{-6}$<br>$7.4\times10^{-7}$ | | Yaws (2003)<br>Gharagheizi et al. (2012)<br>Gharagheizi et al. (2010) | X<br>Q<br>Q | 237<br><br>246 |
| 2,2,3,4,5,5-hexamethylhexane<br>$C_{12}H_{26}$<br>[62185-14-2]<br>NXZMBOYMQLHOPD-UHFFFAOYSA-N | $6.2\times10^{-7}$<br>$7.0\times10^{-7}$<br>$7.4\times10^{-7}$ | | Yaws (2003)<br>Gharagheizi et al. (2012)<br>Gharagheizi et al. (2010) | X<br>Q<br>Q | 237<br><br>246 |



Table A2.1: Alkanes (...continued)

| Substance Formula (Trivial Name) [CAS Registry Number] InChIKey | $H_s^{cp}$ (at $T^{\ominus}$) $\left[\dfrac{\text{mol}}{\text{m}^3\,\text{Pa}}\right]$ | $\dfrac{\text{d}\ln H_s^{cp}}{\text{d}(1/T)}$ [K] | Reference | Type | Note |
|---|---|---|---|---|---|
| 2,3,3,4,4,5-hexamethylhexane C$_{12}$H$_{26}$ [52670-36-7] GRVNDHRZJGSRDC-UHFFFAOYSA-N | $5.6\times10^{-7}$ $1.1\times10^{-6}$ $7.4\times10^{-7}$ | | Yaws (2003) Gharagheizi et al. (2012) Gharagheizi et al. (2010) | X Q Q | 237 246 |
| 2,2,3,3-tetramethyl-4-ethylhexane C$_{12}$H$_{26}$ [62184-98-9] CRTTYRRLFPCOTC-UHFFFAOYSA-N | $6.3\times10^{-7}$ $9.6\times10^{-7}$ $7.2\times10^{-7}$ | | Yaws (2003) Gharagheizi et al. (2012) Gharagheizi et al. (2010) | X Q Q | 237 246 |
| 2,2,3,4-tetramethyl-3-ethylhexane C$_{12}$H$_{26}$ [62184-99-0] SYWRGACXWPDDMM-UHFFFAOYSA-N | $6.2\times10^{-7}$ $1.0\times10^{-6}$ $7.2\times10^{-7}$ | | Yaws (2003) Gharagheizi et al. (2012) Gharagheizi et al. (2010) | X Q Q | 237 246 |
| 2,2,3,4-tetramethyl-4-ethylhexane C$_{12}$H$_{26}$ [62185-00-6] WVLMCSYUMSZJQJ-UHFFFAOYSA-N | $6.1\times10^{-7}$ $1.2\times10^{-6}$ $7.2\times10^{-7}$ | | Yaws (2003) Gharagheizi et al. (2012) Gharagheizi et al. (2010) | X Q Q | 237 246 |
| 2,2,3,5-tetramethyl-3-ethylhexane C$_{12}$H$_{26}$ [62185-01-7] CHURHVBNTYCJDL-UHFFFAOYSA-N | $6.5\times10^{-7}$ $7.9\times10^{-7}$ $7.2\times10^{-7}$ | | Yaws (2003) Gharagheizi et al. (2012) Gharagheizi et al. (2010) | X Q Q | 237 246 |
| 2,2,3,5-tetramethyl-4-ethylhexane C$_{12}$H$_{26}$ [62185-02-8] GYAMSLKQDQYJGT-UHFFFAOYSA-N | $7.4\times10^{-7}$ $7.2\times10^{-7}$ $8.0\times10^{-7}$ | | Yaws (2003) Gharagheizi et al. (2012) Gharagheizi et al. (2010) | X Q Q | 237 246 |
| 2,2,4,4-tetramethyl-3-ethylhexane C$_{12}$H$_{26}$ [62185-03-9] HWHADLCAKCOTRI-UHFFFAOYSA-N | $6.3\times10^{-7}$ $9.3\times10^{-7}$ $7.2\times10^{-7}$ | | Yaws (2003) Gharagheizi et al. (2012) Gharagheizi et al. (2010) | X Q Q | 237 246 |
| 2,2,4,5-tetramethyl-3-ethylhexane C$_{12}$H$_{26}$ [62185-04-0] KZHIDSBQFJFWDV-UHFFFAOYSA-N | $7.4\times10^{-7}$ $7.0\times10^{-7}$ $8.0\times10^{-7}$ | | Yaws (2003) Gharagheizi et al. (2012) Gharagheizi et al. (2010) | X Q Q | 237 246 |
| 2,2,4,5-tetramethyl-4-ethylhexane C$_{12}$H$_{26}$ [62185-05-1] HAVVVIZEMKFOFL-UHFFFAOYSA-N | $6.4\times10^{-7}$ $8.6\times10^{-7}$ $7.2\times10^{-7}$ | | Yaws (2003) Gharagheizi et al. (2012) Gharagheizi et al. (2010) | X Q Q | 237 246 |
| 2,2,5,5-tetramethyl-3-ethylhexane C$_{12}$H$_{26}$ [62185-06-2] GJJUHGLRQQDIFN-UHFFFAOYSA-N | $7.5\times10^{-7}$ $5.2\times10^{-7}$ $7.2\times10^{-7}$ | | Yaws (2003) Gharagheizi et al. (2012) Gharagheizi et al. (2010) | X Q Q | 237 246 |



Table A2.1: Alkanes (. . . continued)

| Substance<br>Formula<br>(Trivial Name)<br>[CAS Registry Number]<br>InChIKey | $H_s^{cp}$<br>(at $T^\ominus$)<br>$\left[\dfrac{\mathrm{mol}}{\mathrm{m^3\,Pa}}\right]$ | $\dfrac{\mathrm{d}\ln H_s^{cp}}{\mathrm{d}(1/T)}$<br><br>[K] | Reference | Type | Note |
|---|---|---|---|---|---|
| 2,3,3,4-tetramethyl-4-ethylhexane<br>$C_{12}H_{26}$<br>[62185-07-3]<br>YVKQGHGRZXUTNK-UHFFFAOYSA-N | $6.1\times10^{-7}$<br>$1.2\times10^{-6}$<br>$7.2\times10^{-7}$ | | Yaws (2003)<br>Gharagheizi et al. (2012)<br>Gharagheizi et al. (2010) | X<br>Q<br>Q | 237<br><br>246 |
| 2,3,3,5-tetramethyl-4-ethylhexane<br>$C_{12}H_{26}$<br>[62185-08-4]<br>AMMSTTMNZHWREI-UHFFFAOYSA-N | $7.1\times10^{-7}$<br>$8.5\times10^{-7}$<br>$8.0\times10^{-7}$ | | Yaws (2003)<br>Gharagheizi et al. (2012)<br>Gharagheizi et al. (2010) | X<br>Q<br>Q | 237<br><br>246 |
| 2,3,4,4-tetramethyl-3-ethylhexane<br>$C_{12}H_{26}$<br>[62185-09-5]<br>VYCKEOSFJUAFGK-UHFFFAOYSA-N | $6.1\times10^{-7}$<br>$1.2\times10^{-6}$<br>$7.2\times10^{-7}$ | | Yaws (2003)<br>Gharagheizi et al. (2012)<br>Gharagheizi et al. (2010) | X<br>Q<br>Q | 237<br><br>246 |
| 2,3,4,5-tetramethyl-3-ethylhexane<br>$C_{12}H_{26}$<br>[62185-10-8]<br>IBPLCPRMAQRHHK-UHFFFAOYSA-N | $6.9\times10^{-7}$<br>$9.6\times10^{-7}$<br>$8.0\times10^{-7}$ | | Yaws (2003)<br>Gharagheizi et al. (2012)<br>Gharagheizi et al. (2010) | X<br>Q<br>Q | 237<br><br>246 |
| 2,2,3-trimethyl-3-isopropylhexane<br>$C_{12}H_{26}$<br>[62199-79-5]<br>UTCSLOLKGOKPGW-UHFFFAOYSA-N | $6.3\times10^{-7}$<br>$9.6\times10^{-7}$<br>$7.2\times10^{-7}$ | | Yaws (2003)<br>Gharagheizi et al. (2012)<br>Gharagheizi et al. (2010) | X<br>Q<br>Q | 237<br><br>246 |
| 2,2,4-trimethyl-3-isopropylhexane<br>$C_{12}H_{26}$<br>[62199-80-8]<br>VEMACPZMOMAUAN-UHFFFAOYSA-N | $7.5\times10^{-7}$<br>$6.8\times10^{-7}$<br>$8.0\times10^{-7}$ | | Yaws (2003)<br>Gharagheizi et al. (2012)<br>Gharagheizi et al. (2010) | X<br>Q<br>Q | 237<br><br>246 |
| 2,2,5-trimethyl-3-isopropylhexane<br>$C_{12}H_{26}$<br>[62199-81-9]<br>GZFAXOCUKPGWEA-UHFFFAOYSA-N | $8.2\times10^{-7}$<br>$5.2\times10^{-7}$<br>$8.0\times10^{-7}$ | | Yaws (2003)<br>Gharagheizi et al. (2012)<br>Gharagheizi et al. (2010) | X<br>Q<br>Q | 237<br><br>246 |
| 2,2,5-trimethyl-4-isopropylhexane<br>$C_{12}H_{26}$<br>[62199-82-0]<br>QFLYTOVYLBZBJC-UHFFFAOYSA-N | $8.2\times10^{-7}$<br>$5.2\times10^{-7}$<br>$8.0\times10^{-7}$ | | Yaws (2003)<br>Gharagheizi et al. (2012)<br>Gharagheizi et al. (2010) | X<br>Q<br>Q | 237<br><br>246 |
| 2,3,4-trimethyl-3-isopropylhexane<br>$C_{12}H_{26}$<br>[62199-83-1]<br>SXELEUFFBZGXIC-UHFFFAOYSA-N | $6.8\times10^{-7}$<br>$1.0\times10^{-6}$<br>$8.0\times10^{-7}$ | | Yaws (2003)<br>Gharagheizi et al. (2012)<br>Gharagheizi et al. (2010) | X<br>Q<br>Q | 237<br><br>246 |
| 2,3,5-trimethyl-3-isopropylhexane<br>$C_{12}H_{26}$<br>[62199-84-2]<br>GUMUKRVQSSIBJO-UHFFFAOYSA-N | $7.2\times10^{-7}$<br>$7.8\times10^{-7}$<br>$8.0\times10^{-7}$ | | Yaws (2003)<br>Gharagheizi et al. (2012)<br>Gharagheizi et al. (2010) | X<br>Q<br>Q | 237<br><br>246 |



Table A2.1: Alkanes (... continued)

| Substance Formula (Trivial Name) [CAS Registry Number] InChIKey | $H_s^{cp}$ (at $T^\ominus$) $\left[\dfrac{\mathrm{mol}}{\mathrm{m^3\,Pa}}\right]$ | $\dfrac{\mathrm{d}\ln H_s^{cp}}{\mathrm{d}(1/T)}$ [K] | Reference | Type | Note |
|---|---|---|---|---|---|
| 2,3,5-trimethyl-4-isopropylhexane $C_{12}H_{26}$ [62199-85-3] VBRDSJSWPGXVIR-UHFFFAOYSA-N | $8.2\times10^{-7}$ $6.9\times10^{-7}$ $9.0\times10^{-7}$ | | Yaws (2003) Gharagheizi et al. (2012) Gharagheizi et al. (2010) | X Q Q | 237 246 |
| 2,4,4-trimethyl-3-isopropylhexane $C_{12}H_{26}$ [62199-86-4] UGPHXQWIHGYHOH-UHFFFAOYSA-N | $7.3\times10^{-7}$ $7.4\times10^{-7}$ $8.0\times10^{-7}$ | | Yaws (2003) Gharagheizi et al. (2012) Gharagheizi et al. (2010) | X Q Q | 237 246 |
| 2,2-dimethyl-3,4-diethylhexane $C_{12}H_{26}$ [62199-89-7] ZAKPEOVUOZWBNO-UHFFFAOYSA-N | $8.1\times10^{-7}$ $6.9\times10^{-7}$ $8.0\times10^{-7}$ | | Yaws (2003) Gharagheizi et al. (2012) Gharagheizi et al. (2010) | X Q Q | 237 246 |
| 2,2-dimethyl-4,4-diethylhexane $C_{12}H_{26}$ [62184-89-8] CYPRMBGQWCLJBN-UHFFFAOYSA-N | $6.9\times10^{-7}$ $9.3\times10^{-7}$ $7.5\times10^{-7}$ | | Yaws (2003) Gharagheizi et al. (2012) Gharagheizi et al. (2010) | X Q Q | 237 246 |
| 2,3-dimethyl-3,4-diethylhexane $C_{12}H_{26}$ [62184-90-1] BUUBMKJJLVCTKU-UHFFFAOYSA-N | $7.5\times10^{-7}$ $1.0\times10^{-6}$ $8.0\times10^{-7}$ | | Yaws (2003) Gharagheizi et al. (2012) Gharagheizi et al. (2010) | X Q Q | 237 246 |
| 2,3-dimethyl-4,4-diethylhexane $C_{12}H_{26}$ [62184-91-2] YNHBDJNMYMZVLM-UHFFFAOYSA-N | $7.5\times10^{-7}$ $1.0\times10^{-6}$ $8.0\times10^{-7}$ | | Yaws (2003) Gharagheizi et al. (2012) Gharagheizi et al. (2010) | X Q Q | 237 246 |
| 2,4-dimethyl-3,3-diethylhexane $C_{12}H_{26}$ [62184-92-3] OXAROVXPKXMTIV-UHFFFAOYSA-N | $7.6\times10^{-7}$ $9.2\times10^{-7}$ $8.0\times10^{-7}$ | | Yaws (2003) Gharagheizi et al. (2012) Gharagheizi et al. (2010) | X Q Q | 237 246 |
| 2,4-dimethyl-3,4-diethylhexane $C_{12}H_{26}$ [62184-93-4] AUVJTUXGNKLTIN-UHFFFAOYSA-N | $7.6\times10^{-7}$ $9.2\times10^{-7}$ $8.0\times10^{-7}$ | | Yaws (2003) Gharagheizi et al. (2012) Gharagheizi et al. (2010) | X Q Q | 237 246 |
| 2,5-dimethyl-3,3-diethylhexane $C_{12}H_{26}$ [62184-94-5] XHDPVTDRZGCNFV-UHFFFAOYSA-N | $8.1\times10^{-7}$ $6.9\times10^{-7}$ $8.0\times10^{-7}$ | | Yaws (2003) Gharagheizi et al. (2012) Gharagheizi et al. (2010) | X Q Q | 237 246 |
| 2,5-dimethyl-3,4-diethylhexane $C_{12}H_{26}$ [62184-95-6] WCPAKUFSQFCWTQ-UHFFFAOYSA-N | $8.9\times10^{-7}$ $6.9\times10^{-7}$ $8.9\times10^{-7}$ | | Yaws (2003) Gharagheizi et al. (2012) Gharagheizi et al. (2010) | X Q Q | 237 246 |



Table A2.1: Alkanes (... continued)

| Substance Formula (Trivial Name) [CAS Registry Number] InChIKey | $H_s^{cp}$ (at $T^{\ominus}$) $\left[ \dfrac{\text{mol}}{\text{m}^3\,\text{Pa}} \right]$ | $\dfrac{\text{d}\ln H_s^{cp}}{\text{d}(1/T)}$ [K] | Reference | Type | Note |
|---|---|---|---|---|---|
| 3,3-dimethyl-4,4-diethylhexane $C_{12}H_{26}$ [62184-96-7] XFUAJHJJXMVZCQ-UHFFFAOYSA-N | $6.7\times10^{-7}$ $1.2\times10^{-6}$ $7.5\times10^{-7}$ | | Yaws (2003) Gharagheizi et al. (2012) Gharagheizi et al. (2010) | X Q Q | 237 246 |
| 3,4-dimethyl-3,4-diethylhexane $C_{12}H_{26}$ [62184-97-8] NTQKWVJMJSYRDX-UHFFFAOYSA-N | $6.7\times10^{-7}$ $1.2\times10^{-6}$ $7.5\times10^{-7}$ | | Yaws (2003) Gharagheizi et al. (2012) Gharagheizi et al. (2010) | X Q Q | 237 246 |
| 2,2-dimethyl-3-*tert*-butylhexane $C_{12}H_{26}$ [62199-76-2] VZAYJUSYQVNGBC-UHFFFAOYSA-N | $6.4\times10^{-7}$ $8.3\times10^{-7}$ $7.2\times10^{-7}$ | | Yaws (2003) Gharagheizi et al. (2012) Gharagheizi et al. (2010) | X Q Q | 237 246 |
| 2-methyl-3-ethyl-3-isopropylhexane $C_{12}H_{26}$ [62199-77-3] TXYYDQJNRQJTOW-UHFFFAOYSA-N | $8.3\times10^{-7}$ $9.2\times10^{-7}$ $8.0\times10^{-7}$ | | Yaws (2003) Gharagheizi et al. (2012) Gharagheizi et al. (2010) | X Q Q | 237 246 |
| 2-methyl-4-ethyl-3-isopropylhexane $C_{12}H_{26}$ [62199-78-4] JXHSGKIMXDNKIS-UHFFFAOYSA-N | $9.7\times10^{-7}$ $6.8\times10^{-7}$ $8.9\times10^{-7}$ | | Yaws (2003) Gharagheizi et al. (2012) Gharagheizi et al. (2010) | X Q Q | 237 246 |
| 3,3,4-triethylhexane $C_{12}H_{26}$ [62199-87-5] YGNQREHYYJPYLX-UHFFFAOYSA-N | $8.2\times10^{-7}$ $1.1\times10^{-6}$ $8.7\times10^{-7}$ | | Yaws (2003) Gharagheizi et al. (2012) Gharagheizi et al. (2010) | X Q Q | 237 246 |
| 2,2,3,4,4-pentamethyl-3-ethylpentane $C_{12}H_{26}$ [66576-21-4] IZNAUALMSCRPBS-UHFFFAOYSA-N | $4.9\times10^{-7}$ $1.4\times10^{-6}$ $6.7\times10^{-7}$ | | Yaws (2003) Gharagheizi et al. (2012) Gharagheizi et al. (2010) | X Q Q | 237 246 |
| 2,2,3,4-tetramethyl-3-isopropylpentane $C_{12}H_{26}$ [62185-17-5] OKGFCJOAXUYCBG-UHFFFAOYSA-N | $5.6\times10^{-7}$ $1.1\times10^{-6}$ $7.4\times10^{-7}$ | | Yaws (2003) Gharagheizi et al. (2012) Gharagheizi et al. (2010) | X Q Q | 237 246 |
| 2,2,4,4-tetramethyl-3-isopropylpentane $C_{12}H_{26}$ [62185-18-6] BYIFHEDQKBAFSW-UHFFFAOYSA-N | $6.0\times10^{-7}$ $7.9\times10^{-7}$ $7.4\times10^{-7}$ | | Yaws (2003) Gharagheizi et al. (2012) Gharagheizi et al. (2010) | X Q Q | 237 246 |



Table A2.1: Alkanes (...continued)

| Substance<br>Formula<br>(Trivial Name)<br>[CAS Registry Number]<br>InChIKey | $H_s^{cp}$<br>(at $T^\ominus$)<br>$\left[\dfrac{\mathrm{mol}}{\mathrm{m}^3\,\mathrm{Pa}}\right]$ | $\dfrac{\mathrm{d}\ln H_s^{cp}}{\mathrm{d}(1/T)}$<br><br>[K] | Reference | Type | Note |
|---|---|---|---|---|---|
| 2,2,4-trimethyl-3,3-diethylpentane<br>$C_{12}H_{26}$<br>[62185-15-3]<br>GIKXQMPMOJVRDN-UHFFFAOYSA-N | $6.1\times10^{-7}$<br>$1.1\times10^{-6}$<br>$7.2\times10^{-7}$ | | Yaws (2003)<br>Gharagheizi et al. (2012)<br>Gharagheizi et al. (2010) | X<br>Q<br>Q | 237<br><br>246 |
| 2,4-dimethyl-3-ethyl-3-isopropylpentane<br>$C_{12}H_{26}$<br>[62185-16-4]<br>ASDDXPPGGPFNIL-UHFFFAOYSA-N | $6.8\times10^{-7}$<br><br>$1.1\times10^{-6}$<br>$8.0\times10^{-7}$ | | Yaws (2003)<br><br>Gharagheizi et al. (2012)<br>Gharagheizi et al. (2010) | X<br><br>Q<br>Q | 237<br><br><br>246 |
| tridecane<br>$C_{13}H_{28}$<br>[629-50-5]<br>IIYFAKIEWZDVMP-UHFFFAOYSA-N | $3.4\times10^{-6}$<br>$1.4\times10^{-6}$<br>$4.4\times10^{-4}$<br>$5.9\times10^{-6}$<br>$2.2\times10^{-6}$<br>$7.9\times10^{-7}$<br>$1.5\times10^{-6}$<br>$4.3\times10^{-6}$ | | Duchowicz et al. (2020)<br>Yaws (2003)<br>Duchowicz et al. (2020)<br>Gharagheizi et al. (2012)<br>Gharagheizi et al. (2010)<br>Hilal et al. (2008)<br>Yaws (1999)<br>Yaws and Yang (1992) | V<br>X<br>Q<br>Q<br>Q<br>Q<br>?<br>? | 186<br>237<br><br><br>246<br><br>21<br>21 |
| 2-methyldodecane<br>$C_{13}H_{28}$<br>[1560-97-0]<br>HGEMCUOAMCILCP-UHFFFAOYSA-N | $1.5\times10^{-6}$<br>$4.8\times10^{-6}$<br>$1.6\times10^{-6}$ | | Yaws (2003)<br>Gharagheizi et al. (2012)<br>Gharagheizi et al. (2010) | X<br>Q<br>Q | 237<br><br>246 |
| 3-methyldodecane<br>$C_{13}H_{28}$<br>[17312-57-1]<br>GRJUENNHVNYCHD-UHFFFAOYSA-N | $1.5\times10^{-6}$<br>$4.9\times10^{-6}$<br>$1.6\times10^{-6}$ | | Yaws (2003)<br>Gharagheizi et al. (2012)<br>Gharagheizi et al. (2010) | X<br>Q<br>Q | 237<br><br>246 |
| 2,2-dimethylundecane<br>$C_{13}H_{28}$<br>[17312-64-0]<br>QDKSGHXRHXVMPF-UHFFFAOYSA-N | $1.1\times10^{-6}$<br>$3.8\times10^{-6}$<br>$1.2\times10^{-6}$ | | Yaws (2003)<br>Gharagheizi et al. (2012)<br>Gharagheizi et al. (2010) | X<br>Q<br>Q | 237<br><br>246 |
| 2,3-dimethylundecane<br>$C_{13}H_{28}$<br>[17312-77-5]<br>QSSUTSOGIQHRIU-UHFFFAOYSA-N | $1.3\times10^{-6}$<br>$4.6\times10^{-6}$<br>$1.2\times10^{-6}$ | | Yaws (2003)<br>Gharagheizi et al. (2012)<br>Gharagheizi et al. (2010) | X<br>Q<br>Q | 237<br><br>246 |
| 2,4-dimethylundecane<br>$C_{13}H_{28}$<br>[17312-80-0]<br>WMZNFELFMFOGCC-UHFFFAOYSA-N | $1.2\times10^{-6}$<br>$2.1\times10^{-6}$<br>$1.2\times10^{-6}$ | | Yaws (2003)<br>Gharagheizi et al. (2012)<br>Gharagheizi et al. (2010) | X<br>Q<br>Q | 237<br><br>246 |



Table A2.1: Alkanes (... continued)

| Substance Formula (Trivial Name) [CAS Registry Number] InChIKey | $H_s^{cp}$ (at $T^{\ominus}$) $\left[\dfrac{\text{mol}}{\text{m}^3\,\text{Pa}}\right]$ | $\dfrac{\text{d}\ln H_s^{cp}}{\text{d}(1/T)}$ [K] | Reference | Type | Note |
|---|---|---|---|---|---|
| tetradecane | $9.3\times10^{-7}$ | | Plyasunov and Shock (2000) | L | |
| C$_{14}$H$_{30}$ | $1.1\times10^{-6}$ | | Duchowicz et al. (2020) | V | 186 |
| [629-59-4] | $1.1\times10^{-6}$ | | HSDB (2015) | V | |
| BGHCVCJVXZWKCC-UHFFFAOYSA-N | $2.6\times10^{-6}$ | | Eastcott et al. (1988) | V | |
| | $7.4\times10^{-6}$ | | Abraham (1984) | V | |
| | $7.2\times10^{-6}$ | | Yaws (2003) | X | 237 |
| | $4.4\times10^{-4}$ | | Duchowicz et al. (2020) | Q | |
| | $6.5\times10^{-6}$ | | Gharagheizi et al. (2012) | Q | |
| | $3.7\times10^{-6}$ | | Gharagheizi et al. (2010) | Q | 246 |
| | $5.6\times10^{-7}$ | | Hilal et al. (2008) | Q | |
| | $2.9\times10^{-5}$ | | Yaffe et al. (2003) | Q | 248, 249 |
| | $2.2\times10^{-6}$ | | Yao et al. (2002) | Q | 229 |
| | $7.2\times10^{-6}$ | | Yaws (1999) | ? | 21 |
| | $8.7\times10^{-6}$ | | Yaws and Yang (1992) | ? | 21 |
| 2-methyltridecane | $2.4\times10^{-6}$ | | Yaws (2003) | X | 237 |
| C$_{14}$H$_{30}$ | $5.2\times10^{-6}$ | | Gharagheizi et al. (2012) | Q | |
| [1560-96-9] | $2.4\times10^{-6}$ | | Gharagheizi et al. (2010) | Q | 246 |
| CJBFZKZYIPBBTO-UHFFFAOYSA-N | | | | | |
| 3-methyltridecane | $2.5\times10^{-6}$ | | Yaws (2003) | X | 237 |
| C$_{14}$H$_{30}$ | $5.4\times10^{-6}$ | | Gharagheizi et al. (2012) | Q | |
| [6418-41-3] | $2.4\times10^{-6}$ | | Gharagheizi et al. (2010) | Q | 246 |
| NLHRRMKILFRDGV-UHFFFAOYSA-N | | | | | |
| 2,2-dimethyldodecane | $1.6\times10^{-6}$ | | Yaws (2003) | X | 237 |
| C$_{14}$H$_{30}$ | $4.3\times10^{-6}$ | | Gharagheizi et al. (2012) | Q | |
| [49598-54-1] | $1.8\times10^{-6}$ | | Gharagheizi et al. (2010) | Q | 246 |
| ATWISEHEXAEGKB-UHFFFAOYSA-N | | | | | |
| 2,3-dimethyldodecane | $2.1\times10^{-6}$ | | Yaws (2003) | X | 237 |
| C$_{14}$H$_{30}$ | $5.2\times10^{-6}$ | | Gharagheizi et al. (2012) | Q | |
| [6117-98-2] | $1.7\times10^{-6}$ | | Gharagheizi et al. (2010) | Q | 246 |
| QBIXLGJCVGNDBJ-UHFFFAOYSA-N | | | | | |
| 2,4-dimethyldodecane | $1.6\times10^{-6}$ | | Yaws (2003) | X | 237 |
| C$_{14}$H$_{30}$ | $3.6\times10^{-6}$ | | Gharagheizi et al. (2012) | Q | |
| [6117-99-3] | $1.7\times10^{-6}$ | | Gharagheizi et al. (2010) | Q | 246 |
| AFELDWXNIFIYOC-UHFFFAOYSA-N | | | | | |
| pentadecane | $7.8\times10^{-7}$ | | Duchowicz et al. (2020) | V | 186 |
| C$_{15}$H$_{32}$ | $7.6\times10^{-7}$ | | HSDB (2015) | V | |
| [629-62-9] | $1.1\times10^{-5}$ | | Yaws (2003) | X | 237 |
| YCOZIPAWZNQLMR-UHFFFAOYSA-N | $4.4\times10^{-4}$ | | Duchowicz et al. (2020) | Q | |
| | $7.1\times10^{-6}$ | | Gharagheizi et al. (2012) | Q | |
| | $7.9\times10^{-6}$ | | Gharagheizi et al. (2010) | Q | 246 |
| | $4.0\times10^{-7}$ | | Hilal et al. (2008) | Q | |
| | $1.1\times10^{-5}$ | | Yaws (1999) | ? | 21 |
| | $2.1\times10^{-5}$ | | Yaws and Yang (1992) | ? | 21 |





Table A2.1: Alkanes (...continued)

| Substance / Formula / (Trivial Name) / [CAS Registry Number] / InChIKey | $H_s^{cp}$ (at $T^\ominus$) $\left[\dfrac{\mathrm{mol}}{\mathrm{m}^3\,\mathrm{Pa}}\right]$ | $\dfrac{\mathrm{d}\ln H_s^{cp}}{\mathrm{d}(1/T)}$ [K] | Reference | Type | Note |
|---|---|---|---|---|---|
| 2-methyltetradecane | $4.9\times10^{-6}$ | | Yaws (2003) | X | 237 |
| $C_{15}H_{32}$ | $5.7\times10^{-6}$ | | Gharagheizi et al. (2012) | Q | |
| [1560-95-8] | $4.7\times10^{-6}$ | | Gharagheizi et al. (2010) | Q | 246 |
| KUVMKLCGXIYSNH-UHFFFAOYSA-N | | | | | |
| 3-methyltetradecane | $5.1\times10^{-6}$ | | Yaws (2003) | X | 237 |
| $C_{15}H_{32}$ | $5.9\times10^{-6}$ | | Gharagheizi et al. (2012) | Q | |
| [18435-22-8] | $4.7\times10^{-6}$ | | Gharagheizi et al. (2010) | Q | 246 |
| HXUYUZCPGPKNGS-UHFFFAOYSA-N | | | | | |
| 2,2-dimethyltridecane | $2.9\times10^{-6}$ | | Yaws (2003) | X | 237 |
| $C_{15}H_{32}$ | $4.5\times10^{-6}$ | | Gharagheizi et al. (2012) | Q | |
| [61869-04-3] | $3.3\times10^{-6}$ | | Gharagheizi et al. (2010) | Q | 246 |
| NVEUWWMNWPNXOC-UHFFFAOYSA-N | | | | | |
| 2,3-dimethyltridecane | $4.3\times10^{-6}$ | | Yaws (2003) | X | 237 |
| $C_{15}H_{32}$ | $5.7\times10^{-6}$ | | Gharagheizi et al. (2012) | Q | |
| [18435-20-6] | $3.1\times10^{-6}$ | | Gharagheizi et al. (2010) | Q | 246 |
| SWUJSLXRUPXTQB-UHFFFAOYSA-N | | | | | |
| 2,4-dimethyltridecane | $2.6\times10^{-6}$ | | Yaws (2003) | X | 237 |
| $C_{15}H_{32}$ | $3.8\times10^{-6}$ | | Gharagheizi et al. (2012) | Q | |
| [61868-05-1] | $3.1\times10^{-6}$ | | Gharagheizi et al. (2010) | Q | 246 |
| JDFJCABQSZLDMZ-UHFFFAOYSA-N | | | | | |
| hexadecane | $2.1\times10^{-5}$ | | Duchowicz et al. (2020) | V | 186 |
| $C_{16}H_{34}$ | $2.6\times10^{-6}$ | | Eastcott et al. (1988) | V | |
| [544-76-3] | $2.7\times10^{-5}$ | | Abraham (1984) | V | |
| DCAYPVUWAIABOU-UHFFFAOYSA-N | $2.1\times10^{-5}$ | | Yaws (2003) | X | 237 |
| | $4.4\times10^{-4}$ | | Duchowicz et al. (2020) | Q | |
| | $7.2\times10^{-6}$ | | Gharagheizi et al. (2012) | Q | |
| | $2.2\times10^{-5}$ | | Gharagheizi et al. (2010) | Q | 246 |
| | $2.9\times10^{-7}$ | | Hilal et al. (2008) | Q | |
| | $2.1\times10^{-5}$ | | Yaws (1999) | ? | 21 |
| | $4.3\times10^{-5}$ | | Yaws and Yang (1992) | ? | 21 |
| 2-methylpentadecane | $1.2\times10^{-5}$ | | Yaws (2003) | X | 237 |
| $C_{16}H_{34}$ | $5.9\times10^{-6}$ | | Gharagheizi et al. (2012) | Q | |
| [1560-93-6] | $1.2\times10^{-5}$ | | Gharagheizi et al. (2010) | Q | 246 |
| BANXPJUEBPWEOT-UHFFFAOYSA-N | | | | | |
| 3-methylpentadecane | $1.2\times10^{-5}$ | | Yaws (2003) | X | 237 |
| $C_{16}H_{34}$ | $6.0\times10^{-6}$ | | Gharagheizi et al. (2012) | Q | |
| [2882-96-4] | $1.2\times10^{-5}$ | | Gharagheizi et al. (2010) | Q | 246 |
| FWXKCXJPHSAYMK-UHFFFAOYSA-N | | | | | |
| 2,2-dimethyltetradecane | $6.5\times10^{-6}$ | | Yaws (2003) | X | 237 |
| $C_{16}H_{34}$ | $4.8\times10^{-6}$ | | Gharagheizi et al. (2012) | Q | |
| [59222-86-5] | $7.9\times10^{-6}$ | | Gharagheizi et al. (2010) | Q | 246 |
| VCAMXEBNASZVEZ-UHFFFAOYSA-N | | | | | |



Table A2.1: Alkanes (. . . continued)

| Substance<br>Formula<br>(Trivial Name)<br>[CAS Registry Number]<br>InChIKey | $H_s^{cp}$<br>(at $T^\ominus$)<br>$\left[\dfrac{\mathrm{mol}}{\mathrm{m^3\,Pa}}\right]$ | $\dfrac{\mathrm{d}\ln H_s^{cp}}{\mathrm{d}(1/T)}$<br><br>[K] | Reference | Type | Note |
|---|---|---|---|---|---|
| 2,3-dimethyltetradecane<br>$C_{16}H_{34}$<br>[18435-23-9]<br>ZQRVEWIJXCRTFW-UHFFFAOYSA-N | $1.1\times10^{-5}$<br>$6.0\times10^{-6}$<br>$7.0\times10^{-6}$ | | Yaws (2003)<br>Gharagheizi et al. (2012)<br>Gharagheizi et al. (2010) | X<br>Q<br>Q | 237<br><br>246 |
| 2,4-dimethyltetradecane<br>$C_{16}H_{34}$<br>[61868-06-2]<br>NWTLFDYNCLTSAR-UHFFFAOYSA-N | $5.1\times10^{-6}$<br>$3.8\times10^{-6}$<br>$7.0\times10^{-6}$ | | Yaws (2003)<br>Gharagheizi et al. (2012)<br>Gharagheizi et al. (2010) | X<br>Q<br>Q | 237<br><br>246 |
| heptadecane<br>$C_{17}H_{36}$<br>[629-78-7]<br>NDJKXXJCMXVBJW-UHFFFAOYSA-N | $2.2\times10^{-7}$<br>$2.1\times10^{-4}$<br>$1.8\times10^{-4}$ | | Hilal et al. (2008)<br>Yaws (1999)<br>Yaws and Yang (1992) | Q<br>?<br>? | <br>21<br>21 |
| octadecane<br>$C_{18}H_{38}$<br>[593-45-3]<br>RZJRJXONCZWCBN-UHFFFAOYSA-N | $1.6\times10^{-6}$<br>$7.8\times10^{-4}$<br>$1.5\times10^{-7}$<br>$6.1\times10^{-4}$<br>$3.6\times10^{-4}$<br>$1.1\times10^{-3}$ | | Eastcott et al. (1988)<br>Abraham (1984)<br>Hilal et al. (2008)<br>Yaffe et al. (2003)<br>Yaws (1999)<br>Yaws and Yang (1992) | V<br>V<br>Q<br>Q<br>?<br>? | <br><br><br>248, 249<br>21<br>21 |
| nonadecane<br>$C_{19}H_{40}$<br>[629-92-5]<br>LQERIDTXQFOHKA-UHFFFAOYSA-N | $1.3\times10^{-7}$<br>$9.3\times10^{-4}$<br>$3.4\times10^{-3}$ | | Hilal et al. (2008)<br>Yaws (1999)<br>Yaws and Yang (1992) | Q<br>?<br>? | <br>21<br>21 |
| eicosane<br>$C_{20}H_{42}$<br>[112-95-8]<br>CBFCDTFDPHXCNY-UHFFFAOYSA-N | $5.0\times10^{-6}$<br>$1.4\times10^{-2}$<br>$9.7\times10^{-8}$<br>$2.3\times10^{-3}$<br>$3.0\times10^{-2}$ | | Eastcott et al. (1988)<br>Abraham (1984)<br>Hilal et al. (2008)<br>Yaws (1999)<br>Yaws and Yang (1992) | V<br>V<br>Q<br>?<br>? | <br><br><br>21<br>21 |
| heneicosane<br>$C_{21}H_{44}$<br>[629-94-7]<br>FNAZRRHPUDJQCJ-UHFFFAOYSA-N | $7.3\times10^{-8}$ | | Hilal et al. (2008) | Q | |
| docosane<br>$C_{22}H_{46}$<br>[629-97-0]<br>HOWGUJZVBDQJKV-UHFFFAOYSA-N | $5.4\times10^{-8}$ | | Hilal et al. (2008) | Q | |
| tricosane<br>$C_{23}H_{48}$<br>[638-67-5]<br>FIGVVZUWCLSUEI-UHFFFAOYSA-N | $4.1\times10^{-8}$ | | Hilal et al. (2008) | Q | |



Table A2.1: Alkanes (...continued)

| Substance Formula (Trivial Name) [CAS Registry Number] InChIKey | $H_s^{cp}$ (at $T^\ominus$) $\left[\dfrac{\text{mol}}{\text{m}^3\,\text{Pa}}\right]$ | $\dfrac{\text{d}\ln H_s^{cp}}{\text{d}(1/T)}$ [K] | Reference | Type | Note |
|---|---|---|---|---|---|
| tetracosane $C_{24}H_{50}$ [646-31-1] POOSGDOYLQNASK-UHFFFAOYSA-N | $3.1\times10^{-8}$ | | Hilal et al. (2008) | Q | |
| pentacosane $C_{25}H_{52}$ [629-99-2] YKNWIILGEFFOPE-UHFFFAOYSA-N | $1.5\times10^{-8}$ | | Hilal et al. (2008) | Q | |
| hexacosane $C_{26}H_{54}$ [630-01-3] HMSWAIKSFDFLKN-UHFFFAOYSA-N | $5.0\times10^{-5}$ $1.3\times10^{2}$ $1.1\times10^{-8}$ | | Eastcott et al. (1988) Abraham (1984) Hilal et al. (2008) | V V Q | |
| heptacosane $C_{27}H_{56}$ [593-49-7] BJQWYEJQWHSSCJ-UHFFFAOYSA-N | $7.7\times10^{-9}$ | | Hilal et al. (2008) | Q | |
| octacosane $C_{28}H_{58}$ [630-02-4] ZYURHZPYMFLWSH-UHFFFAOYSA-N | $5.6\times10^{-9}$ | | Hilal et al. (2008) | Q | |
| nonacosane $C_{29}H_{60}$ [630-03-5] IGGUPRCHHJZPBS-UHFFFAOYSA-N | $4.0\times10^{-9}$ | | Hilal et al. (2008) | Q | |
| triacontane $C_{30}H_{62}$ [638-68-6] JXTPJDDICSTXJX-UHFFFAOYSA-N | $2.9\times10^{-9}$ | | Hilal et al. (2008) | Q | |
| dotriacontane $C_{32}H_{66}$ [544-85-4] QHMGJGNTMQDRQA-UHFFFAOYSA-N | $1.5\times10^{-9}$ | | Hilal et al. (2008) | Q | |
| pentatriacontane $C_{35}H_{72}$ [630-07-9] VHQQPFLOGSTQPC-UHFFFAOYSA-N | $5.8\times10^{-10}$ | | Hilal et al. (2008) | Q | |
| hexatriacontane $C_{36}H_{74}$ [630-06-8] YDLYQMBWCWFRAI-UHFFFAOYSA-N | $8.6\times10^{8}$ | | Abraham (1984) | V | |



Table A2.1: Alkanes (...continued)

| Substance Formula (Trivial Name) [CAS Registry Number] InChIKey | $H_s^{cp}$ (at $T^{\ominus}$) $\left[\dfrac{\text{mol}}{\text{m}^3\,\text{Pa}}\right]$ | $\dfrac{\text{d}\ln H_s^{cp}}{\text{d}(1/T)}$ [K] | Reference | Type | Note |
|---|---|---|---|---|---|
| octatriacontane C$_{38}$H$_{78}$ [7194-85-6] BVKCQBBZBGYNOP-UHFFFAOYSA-N | $2.2 \times 10^{-10}$ | | Hilal et al. (2008) | Q | |





## A2.2 Cycloalkanes

Table A2.2: Cycloalkanes

| Substance Formula (Trivial Name) [CAS Registry Number] InChIKey | $H_s^{cp}$ (at $T^\ominus$) $\left[\dfrac{\mathrm{mol}}{\mathrm{m}^3\,\mathrm{Pa}}\right]$ | $\dfrac{\mathrm{d}\ln H_s^{cp}}{\mathrm{d}(1/T)}$ [K] | Reference | Type | Note |
|---|---|---|---|---|---|
| cyclopropane | $1.2\times10^{-4}$ | 2800 | Plyasunov and Shock (2000) | L | |
| $C_3H_6$ | $1.1\times10^{-4}$ | 1600 | Wilhelm et al. (1977) | L | |
| [75-19-4] | $8.1\times10^{-5}$ | | Steward et al. (1973) | L | 14 |
| LVZWSLJZHVFIQJ-UHFFFAOYSA-N | $1.1\times10^{-4}$ | 2300 | Allott et al. (1973) | L | |
| | $7.8\times10^{-5}$ | | Guitart et al. (1989) | M | 14 |
| | $1.1\times10^{-4}$ | 2000 | Saidman et al. (1966) | M | |
| | $1.2\times10^{-5}$ | | Duchowicz et al. (2020) | V | 186 |
| | $1.2\times10^{-5}$ | | HSDB (2015) | V | |
| | $1.3\times10^{-4}$ | | Irmann (1965) | V | |
| | $1.3\times10^{-4}$ | | Yaws (2003) | X | 237, 294 |
| | $1.1\times10^{-4}$ | | Hayer et al. (2022) | Q | 20 |
| | $3.6\times10^{-3}$ | | Duchowicz et al. (2020) | Q | |
| | $2.6\times10^{-5}$ | | Gharagheizi et al. (2012) | Q | |
| | $6.2\times10^{-5}$ | | Raventos-Duran et al. (2010) | Q | 271, 243 |
| | $1.6\times10^{-4}$ | | Raventos-Duran et al. (2010) | Q | 244 |
| | $9.9\times10^{-5}$ | | Raventos-Duran et al. (2010) | Q | 245 |
| | $1.2\times10^{-4}$ | | Gharagheizi et al. (2010) | Q | 246 |
| | $1.4\times10^{-4}$ | | Hilal et al. (2008) | Q | |
| | $2.3\times10^{-5}$ | | Modarresi et al. (2007) | Q | 67 |
| | | 2500 | Kühne et al. (2005) | Q | |
| | $1.2\times10^{-4}$ | | Yaffe et al. (2003) | Q | 248, 249 |
| | $7.3\times10^{-5}$ | | English and Carroll (2001) | Q | 230, 231 |
| | $3.9\times10^{-5}$ | | Katritzky et al. (1998) | Q | |
| | $9.0\times10^{-5}$ | | Nirmalakhandan et al. (1997) | Q | |
| | | 2200 | Kühne et al. (2005) | ? | |
| | $1.3\times10^{-4}$ | | Yaws (1999) | ? | 21, 294 |
| | $1.1\times10^{-4}$ | 1600 | Yaws et al. (1999) | ? | 21 |
| | $1.3\times10^{-4}$ | | Yaws and Yang (1992) | ? | 21, 294 |
| | $1.1\times10^{-4}$ | | Abraham et al. (1990) | ? | |
| cyclobutane | $1.3\times10^{-6}$ | | Hayer et al. (2022) | Q | 20 |
| $C_4H_8$ | $7.0\times10^{-5}$ | | HSDB (2015) | Q | 99 |
| [287-23-0] | | | | | |
| PMPVIKIVABFJJI-UHFFFAOYSA-N | | | | | |
| cyclopentane | $5.8\times10^{-5}$ | 3500 | Brockbank (2013) | L | 1 |
| $C_5H_{10}$ | $5.5\times10^{-5}$ | 3200 | Plyasunov and Shock (2000) | L | |
| [287-92-3] | $5.4\times10^{-5}$ | | Mackay and Shiu (1981) | L | |
| RGSFGYAAUTVSQA-UHFFFAOYSA-N | $6.5\times10^{-5}$ | 3400 | Hansen et al. (1993) | M | 281 |
| | $5.6\times10^{-5}$ | | Mackay et al. (2006a) | V | |
| | $5.2\times10^{-5}$ | | Mackay et al. (1993) | V | |
| | $5.5\times10^{-5}$ | | Hwang et al. (1992) | V | |
| | $5.4\times10^{-5}$ | | Eastcott et al. (1988) | V | |
| | $5.3\times10^{-5}$ | | Hine and Mookerjee (1975) | V | |
| | $5.6\times10^{-5}$ | | McAuliffe (1963) | V | |
| | $5.3\times10^{-5}$ | | Yaws (2003) | X | 258 |



Table A2.2: Cycloalkanes (... continued)

| Substance Formula (Trivial Name) [CAS Registry Number] InChIKey | $H_s^{cp}$ (at $T^{\ominus}$) $\left[\dfrac{\mathrm{mol}}{\mathrm{m^3\,Pa}}\right]$ | $\dfrac{\mathrm{d}\ln H_s^{cp}}{\mathrm{d}(1/T)}$ [K] | Reference | Type | Note |
|---|---|---|---|---|---|
| | $5.2\times10^{-5}$ | | Yaws (2003) | X | 237 |
| | $6.2\times10^{-5}$ | | Dupeux et al. (2022) | Q | 259 |
| | $4.9\times10^{-5}$ | | Keshavarz et al. (2022) | Q | |
| | $3.6\times10^{-3}$ | | Duchowicz et al. (2020) | Q | 184 |
| | $5.2\times10^{-5}$ | | HSDB (2015) | Q | 99 |
| | $3.0\times10^{-5}$ | | Gharagheizi et al. (2012) | Q | |
| | $3.9\times10^{-5}$ | | Raventos-Duran et al. (2010) | Q | 242, 243 |
| | $1.2\times10^{-4}$ | | Raventos-Duran et al. (2010) | Q | 244 |
| | $4.9\times10^{-5}$ | | Raventos-Duran et al. (2010) | Q | 245 |
| | $5.8\times10^{-5}$ | | Gharagheizi et al. (2010) | Q | 246 |
| | $1.1\times10^{-4}$ | | Hilal et al. (2008) | Q | |
| | $1.2\times10^{-5}$ | | Modarresi et al. (2007) | Q | 67 |
| | | 3200 | Kühne et al. (2005) | Q | |
| | $3.1\times10^{-3}$ | | Yaffe et al. (2003) | Q | 248, 249 |
| | $4.8\times10^{-5}$ | | Yao et al. (2002) | Q | 229 |
| | $3.8\times10^{-5}$ | | English and Carroll (2001) | Q | 230, 231 |
| | $3.5\times10^{-5}$ | | Katritzky et al. (1998) | Q | |
| | $4.3\times10^{-5}$ | | Suzuki et al. (1992) | Q | 232 |
| | $5.7\times10^{-5}$ | | Nirmalakhandan and Speece (1988) | Q | |
| | $6.5\times10^{-5}$ | | Duchowicz et al. (2020) | ? | 185, 21 |
| | | 4300 | Kühne et al. (2005) | ? | |
| | $5.3\times10^{-5}$ | | Yaws (1999) | ? | 21 |
| | $5.2\times10^{-5}$ | | Yaws and Yang (1992) | ? | 21 |
| | $5.3\times10^{-5}$ | | Abraham et al. (1990) | ? | |
| | $5.3\times10^{-5}$ | | Abraham (1979) | ? | |
| cyclohexane $C_6H_{12}$ [110-82-7] XDTMQSROBMDMFD-UHFFFAOYSA-N | $5.1\times10^{-5}$ | 3900 | Brockbank (2013) | L | 1, 295 |
| | $5.3\times10^{-5}$ | 4000 | Plyasunov and Shock (2000) | L | |
| | $5.6\times10^{-5}$ | | Mackay and Shiu (1981) | L | |
| | $3.2\times10^{-4}$ | 5400 | Hiatt (2013) | M | |
| | $8.0\times10^{-5}$ | | Helburn et al. (2008) | M | |
| | $5.2\times10^{-5}$ | 4500 | Dewulf et al. (1999) | M | 296 |
| | $6.0\times10^{-5}$ | | Hansen et al. (1993) | M | 297 |
| | $5.4\times10^{-5}$ | 3800 | Kolb et al. (1992) | M | 277 |
| | $3.4\times10^{-5}$ | | Guitart et al. (1989) | M | 14 |
| | $5.5\times10^{-5}$ | 3200 | Ashworth et al. (1988) | M | 278 |
| | $5.4\times10^{-5}$ | 3400 | Tsonopoulos and Wilson (1983) | M | 1 |
| | $5.4\times10^{-5}$ | 3800 | Tucker et al. (1981) | M | |
| | $5.3\times10^{-5}$ | | Mackay et al. (2006a) | V | |
| | $5.1\times10^{-5}$ | | Mackay et al. (1993) | V | |
| | $6.0\times10^{-5}$ | | Hwang et al. (1992) | V | |
| | $5.4\times10^{-5}$ | | Eastcott et al. (1988) | V | |
| | $5.1\times10^{-5}$ | | Hine and Mookerjee (1975) | V | |
| | $5.6\times10^{-5}$ | | McAuliffe (1963) | V | |
| | $5.4\times10^{-5}$ | 4000 | Plyasunov et al. (2001) | T | |
| | | 4000 | Gill et al. (1976) | T | |
| | $5.0\times10^{-5}$ | | Yaws (2003) | X | 258 |
| | $5.1\times10^{-5}$ | | Yaws (2003) | X | 237 |





Table A2.2: Cycloalkanes (...continued)

| Substance<br>Formula<br>(Trivial Name)<br>[CAS Registry Number]<br>InChIKey | $H_s^{cp}$<br>(at $T^\ominus$)<br><br>$\left[\dfrac{\text{mol}}{\text{m}^3\,\text{Pa}}\right]$ | $\dfrac{\text{d}\ln H_s^{cp}}{\text{d}(1/T)}$<br><br>[K] | Reference | Type | Note |
|---|---|---|---|---|---|
| | $6.2\times10^{-5}$ | 710 | Goldstein (1982) | X | 298 |
| | $6.7\times10^{-5}$ | | Dupeux et al. (2022) | Q | 259 |
| | $6.6\times10^{-5}$ | | Keshavarz et al. (2022) | Q | |
| | $3.6\times10^{-3}$ | | Duchowicz et al. (2020) | Q | 299 |
| | $1.6\times10^{-4}$ | | Wang et al. (2017) | Q | 80, 238 |
| | $9.6\times10^{-5}$ | | Wang et al. (2017) | Q | 80, 239 |
| | $1.5\times10^{-4}$ | | Wang et al. (2017) | Q | 80, 240 |
| | $2.7\times10^{-5}$ | | Gharagheizi et al. (2012) | Q | |
| | $5.0\times10^{-5}$ | | Gharagheizi et al. (2010) | Q | 246 |
| | $9.5\times10^{-5}$ | | Hilal et al. (2008) | Q | |
| | $2.3\times10^{-5}$ | | Modarresi et al. (2007) | Q | 67 |
| | | 3600 | Kühne et al. (2005) | Q | |
| | $5.8\times10^{-5}$ | | Yaffe et al. (2003) | Q | 248, 249 |
| | $3.4\times10^{-5}$ | | Yao et al. (2002) | Q | 229 |
| | $2.7\times10^{-5}$ | | English and Carroll (2001) | Q | 230, 274 |
| | $3.4\times10^{-5}$ | | Katritzky et al. (1998) | Q | |
| | $3.4\times10^{-5}$ | | Suzuki et al. (1992) | Q | 232 |
| | $4.5\times10^{-5}$ | | Nirmalakhandan and Speece (1988) | Q | |
| | $6.6\times10^{-5}$ | | Duchowicz et al. (2020) | ? | 185, 21 |
| | | 3900 | Kühne et al. (2005) | ? | |
| | $5.1\times10^{-5}$ | | Yaws (1999) | ? | 21 |
| | $3.1\times10^{-5}$ | | Abraham and Weathersby (1994) | ? | 21 |
| | $5.1\times10^{-5}$ | | Yaws and Yang (1992) | ? | 21 |
| | $5.1\times10^{-5}$ | | Abraham et al. (1990) | ? | |
| | $5.1\times10^{-5}$ | | Abraham (1979) | ? | |
| methylcyclopentane<br>$C_5H_9CH_3$<br>[96-37-7]<br>GDOPTJXRTPNYNR-UHFFFAOYSA-N | $2.8\times10^{-5}$ | | Brockbank (2013) | L | |
| | $2.8\times10^{-5}$ | 5500 | Plyasunov and Shock (2000) | L | |
| | $2.7\times10^{-5}$ | | Mackay and Shiu (1981) | L | |
| | $2.7\times10^{-5}$ | | Duchowicz et al. (2020) | V | 186 |
| | $2.7\times10^{-5}$ | | HSDB (2015) | V | |
| | $2.8\times10^{-5}$ | | Mackay et al. (2006a) | V | |
| | $2.7\times10^{-5}$ | | Mackay et al. (1993) | V | |
| | $2.7\times10^{-5}$ | | Eastcott et al. (1988) | V | |
| | $2.7\times10^{-5}$ | | Hine and Mookerjee (1975) | V | |
| | $2.9\times10^{-5}$ | | McAuliffe (1963) | V | |
| | $2.8\times10^{-5}$ | | Yaws (2003) | X | 237 |
| | $1.4\times10^{-3}$ | | Duchowicz et al. (2020) | Q | |
| | $1.7\times10^{-5}$ | | Gharagheizi et al. (2012) | Q | |
| | $2.5\times10^{-5}$ | | Raventos-Duran et al. (2010) | Q | 271, 243 |
| | $4.9\times10^{-5}$ | | Raventos-Duran et al. (2010) | Q | 244 |
| | $3.9\times10^{-5}$ | | Raventos-Duran et al. (2010) | Q | 245 |
| | $3.2\times10^{-5}$ | | Gharagheizi et al. (2010) | Q | 246 |
| | $4.4\times10^{-5}$ | | Hilal et al. (2008) | Q | |
| | $1.1\times10^{-5}$ | | Modarresi et al. (2007) | Q | 67 |
| | $2.9\times10^{-5}$ | | Yaffe et al. (2003) | Q | 248, 249 |
| | $2.6\times10^{-5}$ | | Yao et al. (2002) | Q | 229 |
| | $2.7\times10^{-5}$ | | English and Carroll (2001) | Q | 230, 231 |



Table A2.2: Cycloalkanes (. . . continued)

| Substance Formula (Trivial Name) [CAS Registry Number] InChIKey | $H_s^{cp}$ (at $T^{\ominus}$) $\left[\dfrac{\text{mol}}{\text{m}^3\,\text{Pa}}\right]$ | $\dfrac{\text{d}\ln H_s^{cp}}{\text{d}(1/T)}$ [K] | Reference | Type | Note |
|---|---|---|---|---|---|
| | $3.6\times10^{-5}$ | | Katritzky et al. (1998) | Q | |
| | $3.1\times10^{-5}$ | | Suzuki et al. (1992) | Q | 232 |
| | $3.9\times10^{-5}$ | | Nirmalakhandan and Speece (1988) | Q | |
| | $2.8\times10^{-5}$ | | Yaws (1999) | ? | 21 |
| | $1.7\times10^{-5}$ | | Abraham and Weathersby (1994) | ? | 21 |
| | $2.8\times10^{-5}$ | | Yaws and Yang (1992) | ? | 21 |
| cycloheptane $C_7H_{14}$ [291-64-5] DMEGYFMYUHOHGS-UHFFFAOYSA-N | $6.7\times10^{-5}$ | | Brockbank (2013) | L | |
| | $1.0\times10^{-4}$ | 3100 | Plyasunov and Shock (2000) | L | |
| | $1.1\times10^{-4}$ | | Duchowicz et al. (2020) | V | 186 |
| | $8.2\times10^{-5}$ | | Mackay et al. (2006a) | V | |
| | $1.0\times10^{-4}$ | | Mackay et al. (1993) | V | |
| | $1.1\times10^{-4}$ | | Cabani et al. (1981) | V | |
| | $1.1\times10^{-4}$ | | Yaws (2003) | X | 258 |
| | $1.1\times10^{-4}$ | | Yaws (2003) | X | 237 |
| | $1.2\times10^{-4}$ | | Dupeux et al. (2022) | Q | 259 |
| | $3.6\times10^{-3}$ | | Duchowicz et al. (2020) | Q | |
| | $2.5\times10^{-5}$ | | HSDB (2015) | Q | 99 |
| | $3.2\times10^{-5}$ | | Gharagheizi et al. (2012) | Q | |
| | $2.0\times10^{-5}$ | | Raventos-Duran et al. (2010) | Q | 242, 243 |
| | $6.2\times10^{-5}$ | | Raventos-Duran et al. (2010) | Q | 244 |
| | $3.1\times10^{-5}$ | | Raventos-Duran et al. (2010) | Q | 245 |
| | $5.7\times10^{-5}$ | | Gharagheizi et al. (2010) | Q | 246 |
| | $5.1\times10^{-5}$ | | Hilal et al. (2008) | Q | |
| | $8.3\times10^{-6}$ | | Modarresi et al. (2007) | Q | 67 |
| | $1.1\times10^{-4}$ | | Yaffe et al. (2003) | Q | 248, 249 |
| | $3.2\times10^{-5}$ | | Katritzky et al. (1998) | Q | |
| | $1.1\times10^{-4}$ | | Yaws (1999) | ? | 21 |
| | $1.6\times10^{-3}$ | | Hoff et al. (1993) | ? | 21 |
| | $1.1\times10^{-4}$ | | Yaws and Yang (1992) | ? | 21 |
| methylcyclohexane $C_6H_{11}CH_3$ [108-87-2] UAEPNZWRGJTJPN-UHFFFAOYSA-N | $2.6\times10^{-5}$ | 4500 | Brockbank (2013) | L | 1 |
| | $2.4\times10^{-5}$ | | Plyasunov and Shock (2000) | L | |
| | $2.5\times10^{-5}$ | | Mackay and Shiu (1981) | L | |
| | $3.2\times10^{-4}$ | 5300 | Hiatt (2013) | M | |
| | $1.5\times10^{-4}$ | | Ramachandran et al. (1996) | M | |
| | $9.6\times10^{-5}$ | 9400 | Hansen et al. (1993) | M | 281 |
| | $5.0\times10^{-6}$ | | Abraham and Acree (2007) | V | |
| | $2.5\times10^{-5}$ | | Mackay et al. (2006a) | V | |
| | $2.3\times10^{-5}$ | | Mackay et al. (1993) | V | |
| | $2.3\times10^{-5}$ | | Meylan and Howard (1991) | V | |
| | $2.6\times10^{-5}$ | | Eastcott et al. (1988) | V | |
| | $2.3\times10^{-5}$ | | Hine and Mookerjee (1975) | V | |
| | $2.7\times10^{-5}$ | | McAuliffe (1963) | V | |
| | $2.3\times10^{-5}$ | | Yaws (2003) | X | 237 |
| | $8.9\times10^{-5}$ | | Keshavarz et al. (2022) | Q | |
| | $1.4\times10^{-3}$ | | Duchowicz et al. (2020) | Q | 299 |
| | $1.5\times10^{-5}$ | | Gharagheizi et al. (2012) | Q | |



Table A2.2: Cycloalkanes (. . . continued)

| Substance Formula (Trivial Name) [CAS Registry Number] InChIKey | $H_s^{cp}$ (at $T^{\ominus}$) $\left[\dfrac{\mathrm{mol}}{\mathrm{m^3\,Pa}}\right]$ | $\dfrac{\mathrm{d}\ln H_s^{cp}}{\mathrm{d}(1/T)}$ [K] | Reference | Type | Note |
|---|---|---|---|---|---|
| | $2.0\times10^{-5}$ | | Raventos-Duran et al. (2010) | Q | 271, 243 |
| | $3.9\times10^{-5}$ | | Raventos-Duran et al. (2010) | Q | 244 |
| | $3.1\times10^{-5}$ | | Raventos-Duran et al. (2010) | Q | 245 |
| | $2.4\times10^{-5}$ | | Gharagheizi et al. (2010) | Q | 246 |
| | $3.5\times10^{-5}$ | | Hilal et al. (2008) | Q | |
| | $2.1\times10^{-5}$ | | Modarresi et al. (2007) | Q | 67 |
| | | 3900 | Kühne et al. (2005) | Q | |
| | $2.5\times10^{-5}$ | | Yaffe et al. (2003) | Q | 248, 249 |
| | $2.7\times10^{-5}$ | | Yao et al. (2002) | Q | 229 |
| | $1.9\times10^{-5}$ | | English and Carroll (2001) | Q | 230, 260 |
| | $3.5\times10^{-5}$ | | Katritzky et al. (1998) | Q | |
| | $8.6\times10^{-5}$ | | Russell et al. (1992) | Q | 279 |
| | $2.3\times10^{-5}$ | | Suzuki et al. (1992) | Q | 232 |
| | $2.9\times10^{-5}$ | | Meylan and Howard (1991) | Q | |
| | $3.1\times10^{-5}$ | | Nirmalakhandan and Speece (1988) | Q | |
| | $2.3\times10^{-5}$ | | Duchowicz et al. (2020) | ? | 185, 21 |
| | | 3100 | Kühne et al. (2005) | ? | |
| | $2.3\times10^{-5}$ | | Yaws (1999) | ? | 21 |
| | $2.3\times10^{-5}$ | | Yaws and Yang (1992) | ? | 21 |
| methylcyclohexane-d14 $C_6D_{11}CD_3$ [10120-28-2] UAEPNZWRGJTJPN-OBYKGMMLSA-N | $3.1\times10^{-4}$ | 5600 | Hiatt (2013) | M | |
| ethylcyclopentane $C_7H_{14}$ [1640-89-7] IFTRQJLVEBNKJK-UHFFFAOYSA-N | $1.8\times10^{-5}$ $1.6\times10^{-5}$ $1.9\times10^{-5}$ $2.4\times10^{-5}$ $1.8\times10^{-5}$ | | Yaws (2003) Gharagheizi et al. (2012) Gharagheizi et al. (2010) Yao et al. (2002) Yaws (1999) | X Q Q Q ? | 237  246 229 21 |
| 1,1-dimethylcyclopentane $C_7H_{14}$ [1638-26-2] QWHNJUXXYKPLQM-UHFFFAOYSA-N | $2.3\times10^{-5}$ $1.0\times10^{-5}$ $1.9\times10^{-5}$ $1.7\times10^{-5}$ $2.3\times10^{-5}$ | | Yaws (2003) Gharagheizi et al. (2012) Gharagheizi et al. (2010) Yao et al. (2002) Yaws (1999) | X Q Q Q ? | 237  246 229 21 |
| cis-1,2-dimethylcyclopentane $C_7H_{14}$ [1192-18-3] RIRARCHMRDHZAR-KNVOCYPGSA-N | $1.9\times10^{-5}$ $1.4\times10^{-5}$ $1.9\times10^{-5}$ $1.9\times10^{-5}$ | | Yaws (2003) Gharagheizi et al. (2012) Gharagheizi et al. (2010) Yaws (1999) | X Q Q ? | 237  246 21 |
| cis-1,3-dimethylcyclopentane $C_7H_{14}$ [2532-58-3] XAZKFISIRYLAEE-KNVOCYPGSA-N | $2.2\times10^{-5}$ $1.1\times10^{-5}$ $1.9\times10^{-5}$ $2.2\times10^{-5}$ | | Yaws (2003) Gharagheizi et al. (2012) Gharagheizi et al. (2010) Yaws (1999) | X Q Q ? | 237  246 21 |



Table A2.2: Cycloalkanes (. . . continued)

| Substance Formula (Trivial Name) [CAS Registry Number] InChIKey | $H_s^{cp}$ (at $T^\ominus$) $\left[\dfrac{\mathrm{mol}}{\mathrm{m^3\,Pa}}\right]$ | $\dfrac{\mathrm{d}\ln H_s^{cp}}{\mathrm{d}(1/T)}$ [K] | Reference | Type | Note |
|---|---|---|---|---|---|
| *trans*-1,2-dimethylcyclopentane | $2.2\times10^{-5}$ | | Yaws (2003) | X | 237 |
| C$_7$H$_{14}$ | $1.1\times10^{-5}$ | | Gharagheizi et al. (2012) | Q | |
| [822-50-4] | $1.9\times10^{-5}$ | | Gharagheizi et al. (2010) | Q | 246 |
| RIRARCHMRDHZAR-RNFRBKRXSA-N | $2.2\times10^{-5}$ | | Yaws (1999) | ? | 21 |
| *trans*-1,3-dimethylcyclopentane | $2.2\times10^{-5}$ | | Yaws (2003) | X | 237 |
| C$_7$H$_{14}$ | $1.1\times10^{-5}$ | | Gharagheizi et al. (2012) | Q | |
| [1759-58-6] | $1.9\times10^{-5}$ | | Gharagheizi et al. (2010) | Q | 246 |
| XAZKFISIRYLAEE-RNFRBKRXSA-N | $2.2\times10^{-5}$ | | Yaws (1999) | ? | 21 |
| cyclooctane | $7.3\times10^{-5}$ | 4800 | Brockbank (2013) | L | 1 |
| C$_8$H$_{16}$ | $9.3\times10^{-5}$ | 3800 | Plyasunov and Shock (2000) | L | |
| [292-64-8] | $6.9\times10^{-5}$ | 4700 | Dohányosová et al. (2004) | M | 300 |
| WJTCGQSWYFHTAC-UHFFFAOYSA-N | $6.9\times10^{-5}$ | | Mackay et al. (2006a) | V | |
| | $9.3\times10^{-5}$ | | Mackay et al. (1993) | V | |
| | $9.5\times10^{-5}$ | | Cabani et al. (1981) | V | |
| | $9.7\times10^{-5}$ | | Yaws (2003) | X | 237 |
| | $3.1\times10^{-5}$ | | Gharagheizi et al. (2012) | Q | |
| | $9.5\times10^{-5}$ | | Gharagheizi et al. (2010) | Q | 246 |
| | $7.5\times10^{-5}$ | | Hilal et al. (2008) | Q | |
| | $7.4\times10^{-6}$ | | Modarresi et al. (2007) | Q | 67 |
| | | 4300 | Kühne et al. (2005) | Q | |
| | | 5000 | Kühne et al. (2005) | ? | |
| | $9.8\times10^{-5}$ | | Hoff et al. (1993) | ? | 21 |
| | $9.5\times10^{-5}$ | | Yaws and Yang (1992) | ? | 21 |
| methylcycloheptane | $2.1\times10^{-5}$ | | Hilal et al. (2008) | Q | |
| C$_8$H$_{16}$ | | | | | |
| [4126-78-7] | | | | | |
| GYNNXHKOJHMOHS-UHFFFAOYSA-N | | | | | |
| 1,1-dimethylcyclohexane | $1.8\times10^{-5}$ | | Yaws (2003) | X | 237 |
| C$_8$H$_{16}$ | $8.5\times10^{-6}$ | | Gharagheizi et al. (2012) | Q | |
| [590-66-9] | $1.6\times10^{-5}$ | | Gharagheizi et al. (2010) | Q | 246 |
| QEGNUYASOUJEHD-UHFFFAOYSA-N | $2.0\times10^{-5}$ | | Yao et al. (2002) | Q | 229 |
| | $1.8\times10^{-5}$ | | Yaws (1999) | ? | 21 |
| 1,2-dimethylcyclohexane | $2.8\times10^{-5}$ | | Duchowicz et al. (2020) | V | 186 |
| C$_6$H$_{10}$(CH$_3$)$_2$ | $2.1\times10^{-5}$ | | Mackay et al. (1993) | V | |
| [583-57-3] | $5.5\times10^{-4}$ | | Duchowicz et al. (2020) | Q | |
| KVZJLSYJROEPSQ-UHFFFAOYSA-N | $1.6\times10^{-5}$ | | Raventos-Duran et al. (2010) | Q | 271, 243 |
| | $1.6\times10^{-5}$ | | Raventos-Duran et al. (2010) | Q | 244 |
| | $2.0\times10^{-5}$ | | Raventos-Duran et al. (2010) | Q | 245 |
| | $1.4\times10^{-5}$ | | Hilal et al. (2008) | Q | |
| | $3.1\times10^{-5}$ | | Katritzky et al. (1998) | Q | |
| | $1.7\times10^{-5}$ | | Suzuki et al. (1992) | Q | 232 |
| | $2.3\times10^{-5}$ | | Nirmalakhandan and Speece (1988) | Q | |





Table A2.2: Cycloalkanes (. . . continued)

| Substance Formula (Trivial Name) [CAS Registry Number] InChIKey | $H_s^{cp}$ (at $T^{\ominus}$) $\left[\dfrac{\text{mol}}{\text{m}^3\,\text{Pa}}\right]$ | $\dfrac{\text{d}\ln H_s^{cp}}{\text{d}(1/T)}$ [K] | Reference | Type | Note |
|---|---|---|---|---|---|
| *cis*-1,2-dimethylcyclohexane | $2.6\times10^{-5}$ | 4600 | Brockbank (2013) | L | 1 |
| $C_6H_{10}(CH_3)_2$ | $2.8\times10^{-5}$ | | Plyasunov and Shock (2000) | L | |
| [2207-01-4] | $2.8\times10^{-5}$ | | Mackay and Shiu (1981) | L | |
| KVZJLSYJROEPSQ-OCAPTIKFSA-N | $2.7\times10^{-5}$ | 4600 | Dohányosová et al. (2004) | M | 301 |
| | $2.8\times10^{-5}$ | | Duchowicz et al. (2020) | V | 186 |
| | $4.6\times10^{-6}$ | | Abraham and Acree (2007) | V | |
| | $2.8\times10^{-5}$ | | Mackay et al. (2006a) | V | |
| | $2.8\times10^{-5}$ | | Meylan and Howard (1991) | V | |
| | $2.8\times10^{-5}$ | | Eastcott et al. (1988) | V | |
| | $2.8\times10^{-5}$ | | Hine and Mookerjee (1975) | V | |
| | $2.8\times10^{-5}$ | | Yaws (2003) | X | 237 |
| | $5.5\times10^{-4}$ | | Duchowicz et al. (2020) | Q | |
| | $1.1\times10^{-5}$ | | Gharagheizi et al. (2012) | Q | |
| | $1.4\times10^{-5}$ | | Gharagheizi et al. (2010) | Q | 246 |
| | $2.3\times10^{-5}$ | | Modarresi et al. (2007) | Q | 67 |
| | | 4300 | Kühne et al. (2005) | Q | |
| | $2.9\times10^{-5}$ | | Yaffe et al. (2003) | Q | 248, 249 |
| | $1.4\times10^{-5}$ | | English and Carroll (2001) | Q | 230, 231 |
| | $4.3\times10^{-5}$ | | Nirmalakhandan et al. (1997) | Q | |
| | $2.2\times10^{-5}$ | | Meylan and Howard (1991) | Q | |
| | | 4900 | Kühne et al. (2005) | ? | |
| | $2.8\times10^{-5}$ | | Yaws (1999) | ? | 21 |
| | $2.8\times10^{-5}$ | | Yaws and Yang (1992) | ? | 21 |
| *trans*-1,2-dimethylcyclohexane | $1.6\times10^{-5}$ | 4000 | Brockbank (2013) | L | 1 |
| $C_6H_{10}(CH_3)_2$ | $1.6\times10^{-5}$ | 4300 | Dohányosová et al. (2004) | M | 302 |
| [6876-23-9] | $1.8\times10^{-5}$ | | Duchowicz et al. (2020) | V | 186 |
| KVZJLSYJROEPSQ-HTQZYQBOSA-N | $5.7\times10^{-6}$ | | Abraham and Acree (2007) | V | |
| | $1.3\times10^{-5}$ | | Mackay et al. (1993) | V | |
| | $1.6\times10^{-5}$ | | Yaws (2003) | X | 237 |
| | $5.5\times10^{-4}$ | | Duchowicz et al. (2020) | Q | |
| | $9.5\times10^{-6}$ | | Gharagheizi et al. (2012) | Q | |
| | $1.4\times10^{-5}$ | | Gharagheizi et al. (2010) | Q | 246 |
| | | 4300 | Kühne et al. (2005) | Q | |
| | | 4600 | Kühne et al. (2005) | ? | |
| | $1.6\times10^{-5}$ | | Yaws (1999) | ? | 21 |
| | $2.1\times10^{-5}$ | | Yaws and Yang (1992) | ? | 21 |
| | | | Haynes (2014) | W | 303 |
| *cis*-1,3-dimethylcyclohexane | $1.8\times10^{-5}$ | | Yaws (2003) | X | 237 |
| $C_8H_{16}$ | $8.7\times10^{-6}$ | | Gharagheizi et al. (2012) | Q | |
| [638-04-0] | $1.4\times10^{-5}$ | | Gharagheizi et al. (2010) | Q | 246 |
| SGVUHPSBDNVHKL-OCAPTIKFSA-N | $1.8\times10^{-5}$ | | Yaws (1999) | ? | 21 |
| *trans*-1,3-dimethylcyclohexane | $1.7\times10^{-5}$ | | Yaws (2003) | X | 237 |
| $C_8H_{16}$ | $9.8\times10^{-6}$ | | Gharagheizi et al. (2012) | Q | |
| [2207-03-6] | $1.4\times10^{-5}$ | | Gharagheizi et al. (2010) | Q | 246 |
| SGVUHPSBDNVHKL-HTQZYQBOSA-N | $1.7\times10^{-5}$ | | Yaws (1999) | ? | 21 |





Table A2.2: Cycloalkanes (...continued)

| Substance / Formula / (Trivial Name) / [CAS Registry Number] / InChIKey | $H_s^{cp}$ (at $T^{\ominus}$) $\left[\dfrac{\mathrm{mol}}{\mathrm{m^3\,Pa}}\right]$ | $\dfrac{\mathrm{d}\ln H_s^{cp}}{\mathrm{d}(1/T)}$ [K] | Reference | Type | Note |
|---|---|---|---|---|---|
| 1,4-dimethylcyclohexane $C_6H_{10}(CH_3)_2$ [589-90-2] QRMPKOFEUHIBNM-UHFFFAOYSA-N | $1.5\times10^{-5}$ | | Hilal et al. (2008) | Q | |
| *cis*-1,4-dimethylcyclohexane $C_8H_{16}$ [624-29-3] QRMPKOFEUHIBNM-OCAPTIKFSA-N | $1.7\times10^{-5}$ | | Yaws (2003) | X | 237 |
| | $9.8\times10^{-6}$ | | Gharagheizi et al. (2012) | Q | |
| | $1.4\times10^{-5}$ | | Gharagheizi et al. (2010) | Q | 246 |
| | $1.7\times10^{-5}$ | | Yaws (1999) | ? | 21 |
| *trans*-1,4-dimethylcyclohexane $C_6H_{10}(CH_3)_2$ [2207-04-7] QRMPKOFEUHIBNM-ZKCHVHJHSA-N | $1.1\times10^{-5}$ | | Plyasunov and Shock (2000) | L | |
| | $1.1\times10^{-5}$ | | Mackay and Shiu (1981) | L | |
| | $1.1\times10^{-5}$ | | Duchowicz et al. (2020) | V | 186 |
| | $1.1\times10^{-5}$ | | Mackay et al. (2006a) | V | |
| | $1.1\times10^{-5}$ | | Mackay et al. (1993) | V | |
| | $1.1\times10^{-5}$ | | Eastcott et al. (1988) | V | |
| | $1.1\times10^{-5}$ | | Yaws (2003) | X | 237 |
| | $5.5\times10^{-4}$ | | Duchowicz et al. (2020) | Q | |
| | $8.5\times10^{-6}$ | | Gharagheizi et al. (2012) | Q | |
| | $1.6\times10^{-5}$ | | Raventos-Duran et al. (2010) | Q | 242, 243 |
| | $1.6\times10^{-5}$ | | Raventos-Duran et al. (2010) | Q | 244 |
| | $2.0\times10^{-5}$ | | Raventos-Duran et al. (2010) | Q | 245 |
| | $1.4\times10^{-5}$ | | Gharagheizi et al. (2010) | Q | 246 |
| | $2.9\times10^{-5}$ | | Yaffe et al. (2003) | Q | 248, 272 |
| | $1.4\times10^{-5}$ | | English and Carroll (2001) | Q | 230, 231 |
| | $2.2\times10^{-5}$ | | Nirmalakhandan et al. (1997) | Q | |
| | $1.1\times10^{-5}$ | | Yaws (1999) | ? | 21 |
| | $1.1\times10^{-5}$ | | Yaws and Yang (1992) | ? | 21 |
| ethylcyclohexane $C_8H_{16}$ [1678-91-7] IIEWJVIFRVWJOD-UHFFFAOYSA-N | $2.1\times10^{-5}$ | 4100 | Brockbank (2013) | L | 1, 304 |
| | $2.0\times10^{-5}$ | 4400 | Doháryosová et al. (2004) | M | 305 |
| | $3.1\times10^{-5}$ | 4600 | Heidman et al. (1985) | M | 1 |
| | $3.3\times10^{-5}$ | | Duchowicz et al. (2020) | V | 186 |
| | $7.3\times10^{-6}$ | | Abraham and Acree (2007) | V | |
| | $1.5\times10^{-5}$ | | Yaws (2003) | X | 237 |
| | $1.4\times10^{-3}$ | | Duchowicz et al. (2020) | Q | |
| | $1.3\times10^{-5}$ | | Gharagheizi et al. (2012) | Q | |
| | $1.6\times10^{-5}$ | | Raventos-Duran et al. (2010) | Q | 242, 243 |
| | $3.1\times10^{-5}$ | | Raventos-Duran et al. (2010) | Q | 244 |
| | $2.0\times10^{-5}$ | | Raventos-Duran et al. (2010) | Q | 245 |
| | $1.5\times10^{-5}$ | | Gharagheizi et al. (2010) | Q | 246 |
| | $2.3\times10^{-5}$ | | Hilal et al. (2008) | Q | |
| | | 4300 | Kühne et al. (2005) | Q | |
| | $3.3\times10^{-5}$ | | Yaffe et al. (2003) | Q | 248, 249 |
| | $3.5\times10^{-5}$ | | Yao et al. (2002) | Q | 229, 267 |
| | $3.4\times10^{-5}$ | | Katritzky et al. (1998) | Q | |
| | | 4700 | Kühne et al. (2005) | ? | |
| | $1.5\times10^{-5}$ | | Yaws (1999) | ? | 21 |





Table A2.2: Cycloalkanes (... continued)

| Substance<br>Formula<br>(Trivial Name)<br>[CAS Registry Number]<br>InChIKey | $H_s^{cp}$<br>(at $T^{\ominus}$)<br>$\left[\dfrac{\text{mol}}{\text{m}^3\,\text{Pa}}\right]$ | $\dfrac{\text{d}\ln H_s^{cp}}{\text{d}(1/T)}$<br><br>[K] | Reference | Type | Note |
|---|---|---|---|---|---|
| 1,1,2-trimethylcyclopentane | $1.3\times10^{-5}$ | | Yaws (2003) | X | 237 |
| $C_5H_7(CH_3)_3$ | $8.0\times10^{-6}$ | | Gharagheizi et al. (2012) | Q | |
| [4259-00-1] | $1.2\times10^{-5}$ | | Gharagheizi et al. (2010) | Q | 246 |
| WINCSBAYCULVDU-UHFFFAOYSA-N | $6.9\times10^{-6}$ | | Hilal et al. (2008) | Q | |
| 1,1,3-trimethylcyclopentane | $6.2\times10^{-6}$ | | Plyasunov and Shock (2000) | L | |
| $C_5H_7(CH_3)_3$ | $6.3\times10^{-6}$ | | Mackay and Shiu (1981) | L | |
| [4516-69-2] | $6.3\times10^{-6}$ | | Mackay et al. (2006a) | V | |
| OBKHYUIZSOIEPG-UHFFFAOYSA-N | $6.3\times10^{-6}$ | | Mackay et al. (1993) | V | |
| | $6.3\times10^{-6}$ | | Eastcott et al. (1988) | V | |
| | $1.5\times10^{-5}$ | | Yaws (2003) | X | 237 |
| | $6.2\times10^{-6}$ | | Gharagheizi et al. (2012) | Q | |
| | $1.6\times10^{-5}$ | | Raventos-Duran et al. (2010) | Q | 271, 243 |
| | $6.2\times10^{-6}$ | | Raventos-Duran et al. (2010) | Q | 244 |
| | $2.0\times10^{-5}$ | | Raventos-Duran et al. (2010) | Q | 245 |
| | $1.2\times10^{-5}$ | | Gharagheizi et al. (2010) | Q | 246 |
| | $6.2\times10^{-6}$ | | Yaffe et al. (2003) | Q | 248, 249 |
| 1,*cis*-2,*cis*-3-trimethylcyclopentane | $1.1\times10^{-5}$ | | Yaws (2003) | X | 237 |
| $C_8H_{16}$ | $8.9\times10^{-6}$ | | Gharagheizi et al. (2012) | Q | |
| [2613-69-6] | $1.2\times10^{-5}$ | | Gharagheizi et al. (2010) | Q | 246 |
| VCWNHOPGKQCXIQ-RNLVFQAGSA-N | | | | | |
| 1,*cis*-2,*trans*-3-trimethylcyclopentane | $1.2\times10^{-5}$ | | Yaws (2003) | X | 237 |
| $C_8H_{16}$ | $7.6\times10^{-6}$ | | Gharagheizi et al. (2012) | Q | |
| [15890-40-1] | $1.2\times10^{-5}$ | | Gharagheizi et al. (2010) | Q | 246 |
| VCWNHOPGKQCXIQ-JIGDXULJSA-N | | | | | |
| 1,*trans*-2,*cis*-3-trimethylcyclopentane | $1.4\times10^{-5}$ | | Yaws (2003) | X | 237 |
| $C_8H_{16}$ | $6.2\times10^{-6}$ | | Gharagheizi et al. (2012) | Q | |
| [19374-46-0] | $1.2\times10^{-5}$ | | Gharagheizi et al. (2010) | Q | 246 |
| VCWNHOPGKQCXIQ-RNFRBKRXSA-N | | | | | |
| 1,*cis*-2,*cis*-4-trimethylcyclopentane | $1.3\times10^{-5}$ | | Yaws (2003) | X | 237 |
| $C_8H_{16}$ | $7.5\times10^{-6}$ | | Gharagheizi et al. (2012) | Q | |
| [2613-72-1] | $1.2\times10^{-5}$ | | Gharagheizi et al. (2010) | Q | 246 |
| PNUFYSGVPVMNRN-RNLVFQAGSA-N | | | | | |
| 1,*cis*-2,*trans*-4-trimethylcyclopentane | $1.2\times10^{-5}$ | | Yaws (2003) | X | 237 |
| $C_8H_{16}$ | $7.5\times10^{-6}$ | | Gharagheizi et al. (2012) | Q | |
| [4850-28-6] | $1.2\times10^{-5}$ | | Gharagheizi et al. (2010) | Q | 246 |
| PNUFYSGVPVMNRN-WHUPJOBBSA-N | | | | | |



Table A2.2: Cycloalkanes (. . . continued)

| Substance Formula (Trivial Name) [CAS Registry Number] InChIKey | $H_s^{cp}$ (at $T^\ominus$) $\left[\dfrac{\mathrm{mol}}{\mathrm{m^3\,Pa}}\right]$ | $\dfrac{\mathrm{d}\ln H_s^{cp}}{\mathrm{d}(1/T)}$ [K] | Reference | Type | Note |
|---|---|---|---|---|---|
| 1,*trans*-2,*cis*-4-trimethylcyclopentane | $1.4\times10^{-5}$ | | Yaws (2003) | X | 237 |
| $C_8H_{16}$ | $6.0\times10^{-6}$ | | Gharagheizi et al. (2012) | Q | |
| [16883-48-0] | $1.2\times10^{-5}$ | | Gharagheizi et al. (2010) | Q | 246 |
| PNUFYSGVPVMNRN-HTQZYQBOSA-N | | | | | |
| 1-methyl-1-ethylcyclopentane | $1.1\times10^{-5}$ | | Yaws (2003) | X | 237 |
| $C_8H_{16}$ | $9.5\times10^{-6}$ | | Gharagheizi et al. (2012) | Q | |
| [16747-50-5] | $1.2\times10^{-5}$ | | Gharagheizi et al. (2010) | Q | 246 |
| LETYIFNDQBJGPJ-UHFFFAOYSA-N | $1.5\times10^{-5}$ | | Yao et al. (2002) | Q | 229 |
| | $1.1\times10^{-5}$ | | Yaws (1999) | ? | 21 |
| 1-methyl-*cis*-2-ethylcyclopentane | $1.1\times10^{-5}$ | | Yaws (2003) | X | 237 |
| $C_8H_{16}$ | $1.1\times10^{-5}$ | | Gharagheizi et al. (2012) | Q | |
| [930-89-2] | $1.1\times10^{-5}$ | | Gharagheizi et al. (2010) | Q | 246 |
| BSKOLJVTLRLTHE-SFYZADRCSA-N | | | | | |
| 1-methyl-*trans*-2-ethylcyclopentane | $1.2\times10^{-5}$ | | Yaws (2003) | X | 237 |
| $C_8H_{16}$ | $9.1\times10^{-6}$ | | Gharagheizi et al. (2012) | Q | |
| [930-90-5] | $1.1\times10^{-5}$ | | Gharagheizi et al. (2010) | Q | 246 |
| BSKOLJVTLRLTHE-HTQZYQBOSA-N | | | | | |
| 1-methyl-*cis*-3-ethylcyclopentane | $1.2\times10^{-5}$ | | Yaws (2003) | X | 237 |
| $C_8H_{16}$ | $9.0\times10^{-6}$ | | Gharagheizi et al. (2012) | Q | |
| [2613-66-3] | $1.1\times10^{-5}$ | | Gharagheizi et al. (2010) | Q | 246 |
| PQXAPVOKLYINEI-SFYZADRCSA-N | | | | | |
| 1-methyl-*trans*-3-ethylcyclopentane | $1.2\times10^{-5}$ | | Yaws (2003) | X | 237 |
| $C_8H_{16}$ | $9.0\times10^{-6}$ | | Gharagheizi et al. (2012) | Q | |
| [2613-65-2] | $1.1\times10^{-5}$ | | Gharagheizi et al. (2010) | Q | 246 |
| PQXAPVOKLYINEI-HTQZYQBOSA-N | | | | | |
| propylcyclopentane | $1.1\times10^{-5}$ | | Plyasunov and Shock (2000) | L | |
| $C_5H_9C_3H_7$ | $1.1\times10^{-5}$ | | Mackay and Shiu (1981) | L | |
| [2040-96-2] | $1.1\times10^{-5}$ | | Duchowicz et al. (2020) | V | 186 |
| KDIAMAVWIJYWHN-UHFFFAOYSA-N | $1.1\times10^{-5}$ | | Mackay et al. (2006a) | V | |
| | $1.1\times10^{-5}$ | | Mackay et al. (1993) | V | |
| | $1.1\times10^{-5}$ | | Eastcott et al. (1988) | V | |
| | $1.1\times10^{-5}$ | | Yaws (2003) | X | 237 |
| | $1.4\times10^{-3}$ | | Duchowicz et al. (2020) | Q | |
| | $1.2\times10^{-5}$ | | Gharagheizi et al. (2012) | Q | |
| | $1.6\times10^{-5}$ | | Raventos-Duran et al. (2010) | Q | 242, 243 |
| | $2.5\times10^{-5}$ | | Raventos-Duran et al. (2010) | Q | 244 |
| | $2.0\times10^{-5}$ | | Raventos-Duran et al. (2010) | Q | 245 |
| | $1.2\times10^{-5}$ | | Gharagheizi et al. (2010) | Q | 246 |
| | $2.0\times10^{-5}$ | | Hilal et al. (2008) | Q | |
| | $1.1\times10^{-5}$ | | Yaffe et al. (2003) | Q | 248, 249 |
| | $2.5\times10^{-5}$ | | Yao et al. (2002) | Q | 229 |





Table A2.2: Cycloalkanes (...continued)

| Substance Formula (Trivial Name) [CAS Registry Number] InChIKey | $H_s^{cp}$ (at $T^{\ominus}$) $\left[\dfrac{\text{mol}}{\text{m}^3\,\text{Pa}}\right]$ | $\dfrac{\text{d}\ln H_s^{cp}}{\text{d}(1/T)}$ [K] | Reference | Type | Note |
|---|---|---|---|---|---|
| | $3.2\times10^{-5}$ | | Katritzky et al. (1998) | Q | |
| | $2.5\times10^{-5}$ | | Nirmalakhandan et al. (1997) | Q | |
| | $1.1\times10^{-5}$ | | Yaws (1999) | ? | 21 |
| | $1.1\times10^{-5}$ | | Yaws and Yang (1992) | ? | 21 |
| isopropylcyclopentane $C_8H_{16}$ [3875-51-2] TVSBRLGQVHJIKT-UHFFFAOYSA-N | $1.1\times10^{-5}$ $1.1\times10^{-5}$ $1.3\times10^{-5}$ | | Yaws (2003) Gharagheizi et al. (2012) Gharagheizi et al. (2010) | X Q Q | 237 246 |
| 5,5-dimethylbicyclo[2.1.1]hexane $C_8H_{14}$ (MCM:C8BC) WZXMVFMYGAOEAT-UHFFFAOYSA-N | $3.1\times10^{-4}$ $1.7\times10^{-5}$ $2.1\times10^{-4}$ | | Wang et al. (2017) Wang et al. (2017) Wang et al. (2017) | Q Q Q | 80, 238 80, 239 80, 240 |
| 1,1,2-trimethylcyclohexane $C_9H_{18}$ [7094-26-0] MEBONNVPKOBPEA-UHFFFAOYSA-N | $9.0\times10^{-6}$ $8.0\times10^{-6}$ $9.0\times10^{-6}$ | | Yaws (2003) Gharagheizi et al. (2012) Gharagheizi et al. (2010) | X Q Q | 237 246 |
| 1,1,3-trimethylcyclohexane $C_9H_{18}$ [3073-66-3] PYOLJOJPIPCRDP-UHFFFAOYSA-N | $9.3\times10^{-6}$ $9.5\times10^{-6}$ $9.5\times10^{-6}$ $1.2\times10^{-5}$ $6.2\times10^{-6}$ $9.0\times10^{-6}$ | | Plyasunov and Shock (2000) Mackay et al. (2006a) Mackay et al. (1993) Yaws (2003) Gharagheizi et al. (2012) Gharagheizi et al. (2010) | L V V X Q Q | 237 246 |
| 1,1,4-trimethylcyclohexane $C_9H_{18}$ [7094-27-1] UIWORXHEVNIOJG-UHFFFAOYSA-N | $1.1\times10^{-5}$ $5.9\times10^{-6}$ $9.0\times10^{-6}$ | | Yaws (2003) Gharagheizi et al. (2012) Gharagheizi et al. (2010) | X Q Q | 237 246 |
| 1,*cis*-2,*cis*-3-trimethylcyclohexane $C_9H_{18}$ [1839-88-9] DQTVJLHNWPRPPH-AYMMMOKOSA-N | $5.6\times10^{-6}$ $8.6\times10^{-6}$ $8.8\times10^{-6}$ | | Yaws (2003) Gharagheizi et al. (2012) Gharagheizi et al. (2010) | X Q Q | 237 246 |
| 1,*cis*-2,*trans*-3-trimethylcyclohexane $C_9H_{18}$ [7667-55-2] DQTVJLHNWPRPPH-BRPSZJMVSA-N | $6.2\times10^{-6}$ $8.1\times10^{-6}$ $8.8\times10^{-6}$ | | Yaws (2003) Gharagheizi et al. (2012) Gharagheizi et al. (2010) | X Q Q | 237 246 |
| 1,*cis*-2,*cis*-4-trimethylcyclohexane $C_9H_{18}$ [1678-80-4] VCJPCEVERINRSG-YIZRAAEISA-N | $6.2\times10^{-6}$ $7.5\times10^{-6}$ $8.8\times10^{-6}$ | | Yaws (2003) Gharagheizi et al. (2012) Gharagheizi et al. (2010) | X Q Q | 237 246 |



Table A2.2: Cycloalkanes (…continued)

| Substance Formula (Trivial Name) [CAS Registry Number] InChIKey | $H_s^{cp}$ (at $T^\ominus$) $\left[\dfrac{\mathrm{mol}}{\mathrm{m^3\,Pa}}\right]$ | $\dfrac{\mathrm{d}\ln H_s^{cp}}{\mathrm{d}(1/T)}$ [K] | Reference | Type | Note |
|---|---|---|---|---|---|
| 1,*cis*-2,*trans*-4-trimethylcyclohexane | $6.7\times10^{-6}$ | | Yaws (2003) | X | 237 |
| C$_9$H$_{18}$ | $7.5\times10^{-6}$ | | Gharagheizi et al. (2012) | Q | |
| [7667-58-5] | $8.8\times10^{-6}$ | | Gharagheizi et al. (2010) | Q | 246 |
| VCJPCEVERINRSG-HLTSFMKQSA-N | | | | | |
| *cis*,*cis*-1,3,5-trimethylcyclohexane | $1.7\times10^{-5}$ | | Yaws (2003) | X | 237 |
| C$_9$H$_{18}$ | $5.7\times10^{-6}$ | | Gharagheizi et al. (2012) | Q | |
| [1795-27-3] | $8.8\times10^{-6}$ | | Gharagheizi et al. (2010) | Q | 246 |
| ODNRTOSCFYDTKF-AYMMMOKOSA-N | | | | | |
| *cis*,*trans*-1,3,5-trimethylcyclohexane | $1.6\times10^{-5}$ | | Yaws (2003) | X | 237 |
| C$_9$H$_{18}$ | $6.0\times10^{-6}$ | | Gharagheizi et al. (2012) | Q | |
| [1795-26-2] | $6.1\times10^{-6}$ | | Gharagheizi et al. (2012) | Q | |
| | $8.8\times10^{-6}$ | | Gharagheizi et al. (2010) | Q | 246 |
| ODNRTOSCFYDTKF-FBJIGQNJSA-N | | | | | |
| 1,*trans*-2,*cis*-3-trimethylcyclohexane | $7.8\times10^{-6}$ | | Yaws (2003) | X | 237 |
| C$_9$H$_{18}$ | $6.7\times10^{-6}$ | | Gharagheizi et al. (2012) | Q | |
| [1678-81-5] | $8.8\times10^{-6}$ | | Gharagheizi et al. (2010) | Q | 246 |
| DQTVJLHNWPRPPH-HTQZYQBOSA-N | | | | | |
| 1,*trans*-2,*cis*-4-trimethylcyclohexane | $7.1\times10^{-6}$ | | Yaws (2003) | X | 237 |
| C$_9$H$_{18}$ | $6.9\times10^{-6}$ | | Gharagheizi et al. (2012) | Q | |
| [7667-59-6] | $8.8\times10^{-6}$ | | Gharagheizi et al. (2010) | Q | 246 |
| VCJPCEVERINRSG-DJLDLDEBSA-N | | | | | |
| 1,*trans*-2,*trans*-4-trimethylcyclohexane | $7.8\times10^{-6}$ | | Yaws (2003) | X | 237 |
| C$_9$H$_{18}$ | $6.1\times10^{-6}$ | | Gharagheizi et al. (2012) | Q | |
| [7667-60-9] | $8.8\times10^{-6}$ | | Gharagheizi et al. (2010) | Q | 246 |
| VCJPCEVERINRSG-IWSPIJDZSA-N | | | | | |
| 1-methyl-1-ethylcyclohexane | $8.1\times10^{-6}$ | | Yaws (2003) | X | 237 |
| C$_9$H$_{18}$ | $1.0\times10^{-5}$ | | Gharagheizi et al. (2012) | Q | |
| [4926-90-3] | $1.0\times10^{-5}$ | | Gharagheizi et al. (2010) | Q | 246 |
| YPJRYQGOKHKNKZ-UHFFFAOYSA-N | | | | | |
| 1-methyl-*cis*-2-ethylcyclohexane | $9.7\times10^{-6}$ | | Yaws (2003) | X | 237 |
| C$_9$H$_{18}$ | $1.0\times10^{-5}$ | | Gharagheizi et al. (2012) | Q | |
| [4923-77-7] | $8.0\times10^{-6}$ | | Gharagheizi et al. (2010) | Q | 246 |
| XARGIVYWQPXRTC-BDAKNGLRSA-N | | | | | |
| 1-methyl-*cis*-3-ethylcyclohexane | $1.1\times10^{-5}$ | | Yaws (2003) | X | 237 |
| C$_9$H$_{18}$ | $7.9\times10^{-6}$ | | Gharagheizi et al. (2012) | Q | |
| [19489-10-2] | $8.0\times10^{-6}$ | | Gharagheizi et al. (2010) | Q | 246 |
| UDDVMPHNQKRNNS-BDAKNGLRSA-N | | | | | |





Table A2.2: Cycloalkanes (...continued)

| Substance Formula (Trivial Name) [CAS Registry Number] InChIKey | $H_s^{cp}$ (at $T^{\ominus}$) $\left[\dfrac{\text{mol}}{\text{m}^3\,\text{Pa}}\right]$ | $\dfrac{\text{d}\ln H_s^{cp}}{\text{d}(1/T)}$ [K] | Reference | Type | Note |
|---|---|---|---|---|---|
| 1-methyl-*cis*-4-ethylcyclohexane | $1.2\times10^{-5}$ | | Yaws (2003) | X | 237 |
| $C_9H_{18}$ | $8.6\times10^{-6}$ | | Gharagheizi et al. (2012) | Q | |
| [4926-78-7] | $8.0\times10^{-6}$ | | Gharagheizi et al. (2010) | Q | 246 |
| CYISMTMRBPPERU-DTORHVGOSA-N | | | | | |
| 1-methyl-*trans*-2-ethylcyclohexane | $1.0\times10^{-5}$ | | Yaws (2003) | X | 237 |
| $C_9H_{18}$ | $8.8\times10^{-6}$ | | Gharagheizi et al. (2012) | Q | |
| [4923-78-8] | $8.0\times10^{-6}$ | | Gharagheizi et al. (2010) | Q | 246 |
| XARGIVYWQPXRTC-RKDXNWHRSA-N | | | | | |
| 1-methyl-*trans*-3-ethylcyclohexane | $1.1\times10^{-5}$ | | Yaws (2003) | X | 237 |
| $C_9H_{18}$ | $8.6\times10^{-6}$ | | Gharagheizi et al. (2012) | Q | |
| [4926-76-5] | $8.0\times10^{-6}$ | | Gharagheizi et al. (2010) | Q | 246 |
| UDDVMPHNQKRNNS-RKDXNWHRSA-N | | | | | |
| 1-methyl-*trans*-4-ethylcyclohexane | $8.4\times10^{-6}$ | | Yaws (2003) | X | 237 |
| $C_9H_{18}$ | $7.9\times10^{-6}$ | | Gharagheizi et al. (2012) | Q | |
| [6236-88-0] | $8.0\times10^{-6}$ | | Gharagheizi et al. (2010) | Q | 246 |
| CYISMTMRBPPERU-KYZUINATSA-N | | | | | |
| propylcyclohexane | $9.6\times10^{-6}$ | | Yaws (2003) | X | 237 |
| $C_9H_{18}$ | $1.1\times10^{-5}$ | | Gharagheizi et al. (2012) | Q | |
| [1678-92-8] | $1.1\times10^{-5}$ | | Gharagheizi et al. (2010) | Q | 246 |
| DEDZSLCZHWTGOR-UHFFFAOYSA-N | $4.9\times10^{-5}$ | | Yao et al. (2002) | Q | 229 |
| | $9.6\times10^{-6}$ | | Yaws (1999) | ? | 21 |
| isopropylcyclohexane | $9.4\times10^{-6}$ | | Yaws (2003) | X | 237 |
| $C_9H_{18}$ | $1.0\times10^{-5}$ | | Gharagheizi et al. (2012) | Q | |
| [696-29-7] | $1.1\times10^{-5}$ | | Gharagheizi et al. (2010) | Q | 246 |
| GWESVXSMPKAFAS-UHFFFAOYSA-N | $2.7\times10^{-5}$ | | Yao et al. (2002) | Q | 229, 267 |
| | $9.4\times10^{-6}$ | | Yaws (1999) | ? | 21 |
| 1,1,2,2-tetramethylcyclopentane | $8.0\times10^{-6}$ | | Yaws (2003) | X | 237 |
| $C_9H_{18}$ | $5.8\times10^{-6}$ | | Gharagheizi et al. (2012) | Q | |
| [52688-89-8] | $8.5\times10^{-6}$ | | Gharagheizi et al. (2010) | Q | 246 |
| YXDMSFJDVHXFCV-UHFFFAOYSA-N | | | | | |
| *cis*-1,1,2,3-tetramethylcyclopentane | $7.2\times10^{-6}$ | | Yaws (2003) | X | 237 |
| $C_9H_{18}$ | $5.8\times10^{-6}$ | | Gharagheizi et al. (2012) | Q | |
| CXCBKSYSKZEEJB-YUMQZZPRSA-N | $8.6\times10^{-6}$ | | Gharagheizi et al. (2010) | Q | 246 |
| *trans*-1,1,2,3-tetramethylcyclopentane | $8.4\times10^{-6}$ | | Yaws (2003) | X | 237 |
| $C_9H_{18}$ | $4.6\times10^{-6}$ | | Gharagheizi et al. (2012) | Q | |
| [62016-70-0] | $8.6\times10^{-6}$ | | Gharagheizi et al. (2010) | Q | 246 |
| CXCBKSYSKZEEJB-SFYZADRCSA-N | | | | | |



Table A2.2: Cycloalkanes (...continued)

| Substance Formula (Trivial Name) [CAS Registry Number] InChIKey | $H_s^{cp}$ (at $T^\ominus$) $\left[\dfrac{\mathrm{mol}}{\mathrm{m}^3\,\mathrm{Pa}}\right]$ | $\dfrac{\mathrm{d}\ln H_s^{cp}}{\mathrm{d}(1/T)}$ [K] | Reference | Type | Note |
|---|---|---|---|---|---|
| *cis*-1,1,2,4-tetramethylcyclopentane | $8.4\times10^{-6}$ | | Yaws (2003) | X | 237 |
| C$_9$H$_{18}$ | $4.6\times10^{-6}$ | | Gharagheizi et al. (2012) | Q | |
| [62016-71-1] | $8.6\times10^{-6}$ | | Gharagheizi et al. (2010) | Q | 246 |
| AVBGIJNNMIBMQG-SFYZADRCSA-N | | | | | |
| *trans*-1,1,2,4-tetramethylcyclopentane | $8.4\times10^{-6}$ | | Yaws (2003) | X | 237 |
| C$_9$H$_{18}$ | $4.6\times10^{-6}$ | | Gharagheizi et al. (2012) | Q | |
| AVBGIJNNMIBMQG-YUMQZZPRSA-N | $8.6\times10^{-6}$ | | Gharagheizi et al. (2010) | Q | 246 |
| 1,1,3,3-tetramethylcyclopentane | $1.1\times10^{-5}$ | | Yaws (2003) | X | 237 |
| C$_9$H$_{18}$ | $3.8\times10^{-6}$ | | Gharagheizi et al. (2012) | Q | |
| [50876-33-0] | $8.5\times10^{-6}$ | | Gharagheizi et al. (2010) | Q | 246 |
| YWYCGTZNHWYQBD-UHFFFAOYSA-N | | | | | |
| *cis*-1,1,3,4-tetramethylcyclopentane | $8.0\times10^{-6}$ | | Yaws (2003) | X | 237 |
| C$_9$H$_{18}$ | $5.0\times10^{-6}$ | | Gharagheizi et al. (2012) | Q | |
| [53907-60-1] | $8.6\times10^{-6}$ | | Gharagheizi et al. (2010) | Q | 246 |
| OWHFMVURUNNXMJ-OCAPTIKFSA-N | | | | | |
| *trans*-1,1,3,4-tetramethylcyclopentane | $9.9\times10^{-6}$ | | Yaws (2003) | X | 237 |
| C$_9$H$_{18}$ | $3.6\times10^{-6}$ | | Gharagheizi et al. (2012) | Q | |
| [20309-77-7] | $8.6\times10^{-6}$ | | Gharagheizi et al. (2010) | Q | 246 |
| OWHFMVURUNNXMJ-HTQZYQBOSA-N | | | | | |
| *cis*-1,2,2,3-tetramethylcyclopentane | $7.2\times10^{-6}$ | | Yaws (2003) | X | 237 |
| C$_9$H$_{18}$ | $5.8\times10^{-6}$ | | Gharagheizi et al. (2012) | Q | |
| [18938-68-6] | $8.6\times10^{-6}$ | | Gharagheizi et al. (2010) | Q | 246 |
| DHLYDHNCPUAVHP-OCAPTIKFSA-N | | | | | |
| *trans*-1,2,2,3-tetramethylcyclopentane | $7.2\times10^{-6}$ | | Yaws (2003) | X | 237 |
| C$_9$H$_{18}$ | $5.8\times10^{-6}$ | | Gharagheizi et al. (2012) | Q | |
| DHLYDHNCPUAVHP-YUMQZZPRSA-N | $8.6\times10^{-6}$ | | Gharagheizi et al. (2010) | Q | 246 |
| 1,*cis*-2,*cis*-3,*cis*-4-tetramethylcyclopentane | $6.3\times10^{-6}$ | | Yaws (2003) | X | 237 |
| C$_9$H$_{18}$ | $6.2\times10^{-6}$ | | Gharagheizi et al. (2012) | Q | |
| [2532-65-2] | $7.5\times10^{-6}$ | | Gharagheizi et al. (2010) | Q | 246 |
| INYXDKODFMWKER-FNCVBFRFSA-N | | | | | |
| 1,*cis*-2,*cis*-3,*trans*-4-tetramethylcyclopentane | $7.2\times10^{-6}$ | | Yaws (2003) | X | 237 |
| C$_9$H$_{18}$ | $7.5\times10^{-6}$ | | Gharagheizi et al. (2010) | Q | 246 |
| [2532-69-6] | | | | | |
| INYXDKODFMWKER-BGZDPUMWSA-N | | | | | |



Table A2.2: Cycloalkanes (...continued)

| Substance Formula (Trivial Name) [CAS Registry Number] InChIKey | $H_s^{cp}$ (at $T^\ominus$) $\left[\dfrac{\mathrm{mol}}{\mathrm{m^3\,Pa}}\right]$ | $\dfrac{\mathrm{d}\ln H_s^{cp}}{\mathrm{d}(1/T)}$ [K] | Reference | Type | Note |
|---|---|---|---|---|---|
| 1,*cis*-2,*trans*-3,*cis*-4-tetramethylcyclopentane | $8.2\times10^{-6}$ | | Yaws (2003) | X | 237 |
| $C_9H_{18}$ | $4.2\times10^{-6}$ | | Gharagheizi et al. (2012) | Q | |
| [2532-68-5] | $7.5\times10^{-6}$ | | Gharagheizi et al. (2010) | Q | 246 |
| INYXDKODFMWKER-BZNPZCIMSA-N | | | | | |
| 1,*trans*-2,*cis*-3,*trans*-4-tetramethylcyclopentane | $8.8\times10^{-6}$ | | Yaws (2003) | X | 237 |
| $C_9H_{18}$ | $3.7\times10^{-6}$ | | Gharagheizi et al. (2012) | Q | |
| [2532-67-4] | $7.5\times10^{-6}$ | | Gharagheizi et al. (2010) | Q | 246 |
| INYXDKODFMWKER-HXFLIBJXSA-N | | | | | |
| 1,*trans*-2,*trans*-3,*cis*-4-tetramethylcyclopentane | $8.2\times10^{-6}$ | | Yaws (2003) | X | 237 |
| $C_9H_{18}$ | $4.2\times10^{-6}$ | | Gharagheizi et al. (2012) | Q | |
| [19907-40-5] | $7.5\times10^{-6}$ | | Gharagheizi et al. (2010) | Q | 246 |
| INYXDKODFMWKER-OJOKCITNSA-N | | | | | |
| 1,1-dimethyl-2-ethylcyclopentane | $7.7\times10^{-6}$ | | Yaws (2003) | X | 237 |
| $C_9H_{18}$ | $6.2\times10^{-6}$ | | Gharagheizi et al. (2012) | Q | |
| [54549-80-3] | $6.7\times10^{-6}$ | | Gharagheizi et al. (2010) | Q | 246 |
| RXPIHZJWAFCHEJ-UHFFFAOYSA-N | | | | | |
| 1,1-dimethyl-3-ethylcyclopentane | $8.3\times10^{-6}$ | | Yaws (2003) | X | 237 |
| $C_9H_{18}$ | $5.5\times10^{-6}$ | | Gharagheizi et al. (2012) | Q | |
| [62016-61-9] | $6.7\times10^{-6}$ | | Gharagheizi et al. (2010) | Q | 246 |
| WXHYOGXBESCULU-UHFFFAOYSA-N | | | | | |
| 1,*cis*-2-dimethyl-1-ethylcyclopentane | $6.9\times10^{-6}$ | | Yaws (2003) | X | 237 |
| $C_9H_{18}$ | $7.3\times10^{-6}$ | | Gharagheizi et al. (2012) | Q | |
| [62016-63-1] | $6.7\times10^{-6}$ | | Gharagheizi et al. (2010) | Q | 246 |
| DRWDWIMYUGHJBR-DTWKUNHWSA-N | | | | | |
| 1,*trans*-2-dimethyl-1-ethylcyclopentane | $6.9\times10^{-6}$ | | Yaws (2003) | X | 237 |
| $C_9H_{18}$ | $7.3\times10^{-6}$ | | Gharagheizi et al. (2012) | Q | |
| [62016-62-0] | $6.7\times10^{-6}$ | | Gharagheizi et al. (2010) | Q | 246 |
| DRWDWIMYUGHJBR-RKDXNWHRSA-N | | | | | |
| 1,*cis*-2-dimethyl-*cis*-3-ethylcyclopentane | $6.0\times10^{-6}$ | | Yaws (2003) | X | 237 |
| $C_9H_{18}$ | $8.0\times10^{-6}$ | | Gharagheizi et al. (2012) | Q | |
| UMUGNPFWQJAOJI-HLTSFMKQSA-N | $6.7\times10^{-6}$ | | Gharagheizi et al. (2010) | Q | 246 |
| 1,*trans*-2-dimethyl-*trans*-3-ethylcyclopentane | $6.7\times10^{-6}$ | | Yaws (2003) | X | 237 |
| $C_9H_{18}$ | $6.6\times10^{-6}$ | | Gharagheizi et al. (2012) | Q | |
| UMUGNPFWQJAOJI-HRDYMLBCSA-N | $6.7\times10^{-6}$ | | Gharagheizi et al. (2010) | Q | 246 |





Table A2.2: Cycloalkanes (. . . continued)

| Substance Formula (Trivial Name) [CAS Registry Number] InChIKey | $H_s^{cp}$ (at $T^\ominus$) $\left[\dfrac{\text{mol}}{\text{m}^3\,\text{Pa}}\right]$ | $\dfrac{\text{d}\ln H_s^{cp}}{\text{d}(1/T)}$ [K] | Reference | Type | Note |
|---|---|---|---|---|---|
| 1,*cis*-2-dimethyl-*trans*-3-ethylcyclopentane | $6.7\times10^{-6}$ | | Yaws (2003) | X | 237 |
| C$_9$H$_{18}$ | $6.6\times10^{-6}$ | | Gharagheizi et al. (2012) | Q | |
| UMUGNPFWQJAOJI-IWSPIJDZSA-N | $6.7\times10^{-6}$ | | Gharagheizi et al. (2010) | Q | 246 |
| 1,*trans*-2-dimethyl-*cis*-3-ethylcyclopentane | $7.7\times10^{-6}$ | | Yaws (2003) | X | 237 |
| C$_9$H$_{18}$ | $5.4\times10^{-6}$ | | Gharagheizi et al. (2012) | Q | |
| UMUGNPFWQJAOJI-VGMNWLOBSA-N | $6.7\times10^{-6}$ | | Gharagheizi et al. (2010) | Q | 246 |
| 1,*cis*-2-dimethyl-*cis*-4-ethylcyclopentane | $6.5\times10^{-6}$ | | Yaws (2003) | X | 237 |
| C$_9$H$_{18}$ | $7.0\times10^{-6}$ | | Gharagheizi et al. (2012) | Q | |
| [62016-64-2] | $6.7\times10^{-6}$ | | Gharagheizi et al. (2010) | Q | 246 |
| QKXQNQVXTYSGKS-BRPSZJMVSA-N | | | | | |
| 1,*cis*-2-dimethyl-*trans*-4-ethylcyclopentane | $6.5\times10^{-6}$ | | Yaws (2003) | X | 237 |
| C$_9$H$_{18}$ | $7.0\times10^{-6}$ | | Gharagheizi et al. (2012) | Q | |
| [62016-65-3] | $6.7\times10^{-6}$ | | Gharagheizi et al. (2010) | Q | 246 |
| QKXQNQVXTYSGKS-AYMMMOKOSA-N | | | | | |
| 1,*trans*-2-dimethyl-*cis*-4-ethylcyclopentane | $7.4\times10^{-6}$ | | Yaws (2003) | X | 237 |
| C$_9$H$_{18}$ | $5.7\times10^{-6}$ | | Gharagheizi et al. (2012) | Q | |
| [62016-66-4] | $6.7\times10^{-6}$ | | Gharagheizi et al. (2010) | Q | 246 |
| QKXQNQVXTYSGKS-HTQZYQBOSA-N | | | | | |
| 1,*cis*-3-dimethyl-1-ethylcyclopentane | $8.0\times10^{-6}$ | | Yaws (2003) | X | 237 |
| C$_9$H$_{18}$ | $5.8\times10^{-6}$ | | Gharagheizi et al. (2012) | Q | |
| [62016-68-6] | $6.7\times10^{-6}$ | | Gharagheizi et al. (2010) | Q | 246 |
| OAWOMHRJVJMDLZ-DTWKUNHWSA-N | | | | | |
| 1,*trans*-3-dimethyl-1-ethylcyclopentane | $8.0\times10^{-6}$ | | Yaws (2003) | X | 237 |
| C$_9$H$_{18}$ | $5.8\times10^{-6}$ | | Gharagheizi et al. (2012) | Q | |
| [62016-67-5] | $6.7\times10^{-6}$ | | Gharagheizi et al. (2010) | Q | 246 |
| OAWOMHRJVJMDLZ-RKDXNWHRSA-N | | | | | |
| 1,*cis*-3-dimethyl-*cis*-2-ethylcyclopentane | $6.0\times10^{-6}$ | | Yaws (2003) | X | 237 |
| C$_9$H$_{18}$ | $8.0\times10^{-6}$ | | Gharagheizi et al. (2012) | Q | |
| [19903-00-5] | $6.7\times10^{-6}$ | | Gharagheizi et al. (2010) | Q | 246 |
| JREISGVVJTVFBL-AYMMMOKOSA-N | | | | | |
| 1,*cis*-3-dimethyl-*trans*-2-ethylcyclopentane | $7.8\times10^{-6}$ | | Yaws (2003) | X | 237 |
| C$_9$H$_{18}$ | $5.2\times10^{-6}$ | | Gharagheizi et al. (2012) | Q | |
| [19902-98-8] | $6.7\times10^{-6}$ | | Gharagheizi et al. (2010) | Q | 246 |
| JREISGVVJTVFBL-BRPSZJMVSA-N | | | | | |




Table A2.2: Cycloalkanes (...continued)

| Substance Formula (Trivial Name) [CAS Registry Number] InChIKey | $H_s^{cp}$ (at $T^{\ominus}$) $\left[\dfrac{\text{mol}}{\text{m}^3\,\text{Pa}}\right]$ | $\dfrac{\text{d}\ln H_s^{cp}}{\text{d}(1/T)}$ [K] | Reference | Type | Note |
|---|---|---|---|---|---|
| 1,*trans*-3-dimethyl-*cis*-2-ethylcyclopentane | $6.9\times10^{-6}$ | | Yaws (2003) | X | 237 |
| C$_9$H$_{18}$ | $6.4\times10^{-6}$ | | Gharagheizi et al. (2012) | Q | |
| [19902-99-9] | $6.7\times10^{-6}$ | | Gharagheizi et al. (2010) | Q | 246 |
| JREISGVVJTVFBL-HTQZYQBOSA-N | | | | | |
| 1,*cis*-3-dimethyl-*cis*-4-ethylcyclopentane | $6.7\times10^{-6}$ | | Yaws (2003) | X | 237 |
| C$_9$H$_{18}$ | $6.6\times10^{-6}$ | | Gharagheizi et al. (2012) | Q | |
| VMCXXGFUCWAIIN-YIZRAAEISA-N | $6.7\times10^{-6}$ | | Gharagheizi et al. (2010) | Q | 246 |
| 1,*cis*-3-dimethyl-*trans*-4-ethylcyclopentane | $7.8\times10^{-6}$ | | Yaws (2003) | X | 237 |
| C$_9$H$_{18}$ | $5.2\times10^{-6}$ | | Gharagheizi et al. (2012) | Q | |
| VMCXXGFUCWAIIN-DJLDLDEBSA-N | $6.7\times10^{-6}$ | | Gharagheizi et al. (2010) | Q | 246 |
| 1,*trans*-3-dimethyl-*cis*-4-ethylcyclopentane | $3.5\times10^{-6}$ | | Yaws (2003) | X | 237 |
| C$_9$H$_{18}$ | $2.5\times10^{-5}$ | | Gharagheizi et al. (2012) | Q | |
| VMCXXGFUCWAIIN-CIUDSAMLSA-N | $6.7\times10^{-6}$ | | Gharagheizi et al. (2010) | Q | 246 |
| 1,*trans*-3-dimethyl-*trans*-4-ethylcyclopentane | $6.7\times10^{-6}$ | | Yaws (2003) | X | 237 |
| C$_9$H$_{18}$ | $6.6\times10^{-6}$ | | Gharagheizi et al. (2012) | Q | |
| VMCXXGFUCWAIIN-XHNCKOQMSA-N | $6.7\times10^{-6}$ | | Gharagheizi et al. (2010) | Q | 246 |
| 1-methyl-1-propylcyclopentane | $7.2\times10^{-6}$ | | Yaws (2003) | X | 237 |
| C$_9$H$_{18}$ | $8.3\times10^{-6}$ | | Gharagheizi et al. (2012) | Q | |
| [16631-63-3] | $7.5\times10^{-6}$ | | Gharagheizi et al. (2010) | Q | 246 |
| HICYLMKNNFKEMK-UHFFFAOYSA-N | | | | | |
| 1-methyl-*cis*-2-propylcyclopentane | $6.4\times10^{-6}$ | | Yaws (2003) | X | 237 |
| C$_9$H$_{18}$ | $8.6\times10^{-6}$ | | Gharagheizi et al. (2012) | Q | |
| [932-43-4] | $6.4\times10^{-6}$ | | Gharagheizi et al. (2010) | Q | 246 |
| ADQJFBQXLAAVQA-BDAKNGLRSA-N | | | | | |
| 1-methyl-*trans*-2-propylcyclopentane | $7.1\times10^{-6}$ | | Yaws (2003) | X | 237 |
| C$_9$H$_{18}$ | $7.2\times10^{-6}$ | | Gharagheizi et al. (2012) | Q | |
| [932-44-5] | $6.4\times10^{-6}$ | | Gharagheizi et al. (2010) | Q | 246 |
| ADQJFBQXLAAVQA-RKDXNWHRSA-N | | | | | |
| 1-methyl-*cis*-3-propylcyclopentane | $6.9\times10^{-6}$ | | Yaws (2003) | X | 237 |
| C$_9$H$_{18}$ | $7.6\times10^{-6}$ | | Gharagheizi et al. (2012) | Q | |
| [2443-04-1] | $6.4\times10^{-6}$ | | Gharagheizi et al. (2010) | Q | 246 |
| HRSBIYASWAILIF-BDAKNGLRSA-N | | | | | |




Table A2.2: Cycloalkanes (...continued)

| Substance Formula (Trivial Name) [CAS Registry Number] InChIKey | $H_s^{cp}$ (at $T^\ominus$) $\left[\dfrac{\text{mol}}{\text{m}^3\,\text{Pa}}\right]$ | $\dfrac{\text{d}\ln H_s^{cp}}{\text{d}(1/T)}$ [K] | Reference | Type | Note |
|---|---|---|---|---|---|
| 1-methyl-*trans*-3-propylcyclopentane | $6.9\times10^{-6}$ | | Yaws (2003) | X | 237 |
| C$_9$H$_{18}$ | $7.6\times10^{-6}$ | | Gharagheizi et al. (2012) | Q | |
| [2443-03-0] | $6.4\times10^{-6}$ | | Gharagheizi et al. (2010) | Q | 246 |
| HRSBIYASWAILIF-RKDXNWHRSA-N | | | | | |
| 1-methyl-1-isopropylcyclopentane | $6.5\times10^{-6}$ | | Yaws (2003) | X | 237 |
| C$_9$H$_{18}$ | $8.9\times10^{-6}$ | | Gharagheizi et al. (2012) | Q | |
| [61828-00-0] | $7.5\times10^{-6}$ | | Gharagheizi et al. (2010) | Q | 246 |
| XFMQGDWBNOQLEG-UHFFFAOYSA-N | | | | | |
| 1-methyl-*cis*-2-isopropylcyclopentane | $6.5\times10^{-6}$ | | Yaws (2003) | X | 237 |
| C$_9$H$_{18}$ | $7.7\times10^{-6}$ | | Gharagheizi et al. (2012) | Q | |
| [61868-01-7] | $7.0\times10^{-6}$ | | Gharagheizi et al. (2010) | Q | 246 |
| CGWXYEIWDQDFIU-RKDXNWHRSA-N | | | | | |
| 1-methyl-*trans*-2-isopropylcyclopentane | $7.3\times10^{-6}$ | | Yaws (2003) | X | 237 |
| C$_9$H$_{18}$ | $6.3\times10^{-6}$ | | Gharagheizi et al. (2012) | Q | |
| [61828-01-1] | $7.0\times10^{-6}$ | | Gharagheizi et al. (2010) | Q | 246 |
| CGWXYEIWDQDFIU-BDAKNGLRSA-N | | | | | |
| 1-methyl-*cis*-3-isopropylcyclopentane | $7.2\times10^{-6}$ | | Yaws (2003) | X | 237 |
| C$_9$H$_{18}$ | $6.5\times10^{-6}$ | | Gharagheizi et al. (2012) | Q | |
| [61828-02-2] | $7.0\times10^{-6}$ | | Gharagheizi et al. (2010) | Q | 246 |
| CDTDMKCVKCGRPD-BDAKNGLRSA-N | | | | | |
| 1-methyl-*trans*-3-isopropylcyclopentane | $7.2\times10^{-6}$ | | Yaws (2003) | X | 237 |
| C$_9$H$_{18}$ | $6.5\times10^{-6}$ | | Gharagheizi et al. (2012) | Q | |
| [61828-03-3] | $7.0\times10^{-6}$ | | Gharagheizi et al. (2010) | Q | 246 |
| CDTDMKCVKCGRPD-RKDXNWHRSA-N | | | | | |
| 1,1-diethylcyclopentane | $6.7\times10^{-6}$ | | Yaws (2003) | X | 237 |
| C$_9$H$_{18}$ | $9.4\times10^{-6}$ | | Gharagheizi et al. (2012) | Q | |
| [2721-38-2] | $7.5\times10^{-6}$ | | Gharagheizi et al. (2010) | Q | 246 |
| DPGQSDLGKGLNHC-UHFFFAOYSA-N | | | | | |
| 1,*cis*-2-diethylcyclopentane | $6.3\times10^{-6}$ | | Yaws (2003) | X | 237 |
| C$_9$H$_{18}$ | $8.9\times10^{-6}$ | | Gharagheizi et al. (2012) | Q | |
| [932-39-8] | $6.4\times10^{-6}$ | | Gharagheizi et al. (2010) | Q | 246 |
| JKMYLSLBFNMSFP-DTORHVGOSA-N | | | | | |
| 1,*trans*-2-diethylcyclopentane | $6.9\times10^{-6}$ | | Yaws (2003) | X | 237 |
| C$_9$H$_{18}$ | $7.5\times10^{-6}$ | | Gharagheizi et al. (2012) | Q | |
| [932-40-1] | $6.4\times10^{-6}$ | | Gharagheizi et al. (2010) | Q | 246 |
| JKMYLSLBFNMSFP-RKDXNWHRSA-N | | | | | |





Table A2.2: Cycloalkanes (. . . continued)

| Substance Formula (Trivial Name) [CAS Registry Number] InChIKey | $H_s^{cp}$ (at $T^\ominus$) $\left[\dfrac{\text{mol}}{\text{m}^3\,\text{Pa}}\right]$ | $\dfrac{\mathrm{d}\ln H_s^{cp}}{\mathrm{d}(1/T)}$ [K] | Reference | Type | Note |
|---|---|---|---|---|---|
| 1,*cis*-3-diethylcyclopentane C$_9$H$_{18}$ [62016-59-5] RUUVWUNHERVOAY-DTORHVGOSA-N | $6.7\times10^{-6}$ $8.0\times10^{-6}$ $6.4\times10^{-6}$ | | Yaws (2003) Gharagheizi et al. (2012) Gharagheizi et al. (2010) | X Q Q | 237 246 |
| 1,*trans*-3-diethylcyclopentane C$_9$H$_{18}$ [62016-60-8] RUUVWUNHERVOAY-RKDXNWHRSA-N | $6.7\times10^{-6}$ $8.0\times10^{-6}$ $6.4\times10^{-6}$ | | Yaws (2003) Gharagheizi et al. (2012) Gharagheizi et al. (2010) | X Q Q | 237 246 |
| butylcyclopentane C$_9$H$_{18}$ [2040-95-1] ZAGHKONXGGSVDV-UHFFFAOYSA-N | $6.5\times10^{-6}$ $1.0\times10^{-5}$ $8.7\times10^{-6}$ $6.5\times10^{-6}$ | | Yaws (2003) Gharagheizi et al. (2012) Gharagheizi et al. (2010) Yaws (1999) | X Q Q ? | 237 246 21 |
| isobutylcyclopentane C$_9$H$_{18}$ [3788-32-7] DPUYDFJBHDYVQM-UHFFFAOYSA-N | $7.2\times10^{-6}$ $8.2\times10^{-6}$ $8.5\times10^{-6}$ | | Yaws (2003) Gharagheizi et al. (2012) Gharagheizi et al. (2010) | X Q Q | 237 246 |
| *sec*-butylcyclopentane C$_9$H$_{18}$ [4850-32-2] DCEHVBLXWODXCW-UHFFFAOYSA-N | $6.5\times10^{-6}$ $9.8\times10^{-6}$ $8.5\times10^{-6}$ | | Yaws (2003) Gharagheizi et al. (2012) Gharagheizi et al. (2010) | X Q Q | 237 246 |
| *tert*-butylcyclopentane C$_9$H$_{18}$ [3875-52-3] BFWVYBVSRYIDHI-UHFFFAOYSA-N | $6.7\times10^{-6}$ $7.6\times10^{-6}$ $7.7\times10^{-6}$ | | Yaws (2003) Gharagheizi et al. (2012) Gharagheizi et al. (2010) | X Q Q | 237 246 |
| octahydro-1H-indene C$_9$H$_{16}$ [496-10-6] BNRNAKTVFSZAFA-UHFFFAOYSA-N | $8.8\times10^{-5}$ | | Hilal et al. (2008) | Q | |
| butylcyclohexane C$_{10}$H$_{20}$ [1678-93-9] GGBJHURWWWLEQH-UHFFFAOYSA-N | $7.2\times10^{-6}$ $8.5\times10^{-6}$ $9.3\times10^{-6}$ $7.2\times10^{-6}$ | | Yaws (2003) Gharagheizi et al. (2012) Gharagheizi et al. (2010) Yaws (1999) | X Q Q ? | 237 246 21 |
| pentylcyclopentane C$_{10}$H$_{20}$ [3741-00-2] HPQURZRDYMUHJI-UHFFFAOYSA-N | $5.3\times10^{-6}$ $5.4\times10^{-6}$ $5.4\times10^{-6}$ $5.4\times10^{-6}$ $5.4\times10^{-6}$ $5.2\times10^{-6}$ $8.1\times10^{-6}$ $7.8\times10^{-6}$ $1.2\times10^{-5}$ $1.2\times10^{-5}$ $7.0\times10^{-6}$ | | Plyasunov and Shock (2000) Mackay and Shiu (1981) Mackay et al. (2006a) Mackay et al. (1993) Eastcott et al. (1988) Yaws (2003) Gharagheizi et al. (2012) Raventos-Duran et al. (2010) Raventos-Duran et al. (2010) Raventos-Duran et al. (2010) Gharagheizi et al. (2010) | L L V V V X Q Q Q Q Q | 237 242, 243 244 245 246 |





**158** **Rolf Sander: Compilation of Henry's law constants**

Table A2.2: Cycloalkanes (...continued)

| Substance Formula (Trivial Name) [CAS Registry Number] InChIKey | $H_s^{cp}$ (at $T^\ominus$) $\left[\dfrac{\mathrm{mol}}{\mathrm{m^3\,Pa}}\right]$ | $\dfrac{\mathrm{d}\ln H_s^{cp}}{\mathrm{d}(1/T)}$ [K] | Reference | Type | Note |
|---|---|---|---|---|---|
| | $9.2\times10^{-6}$ | | Hilal et al. (2008) | Q | |
| | $5.8\times10^{-6}$ | | Yaffe et al. (2003) | Q | 248, 249 |
| | $1.6\times10^{-5}$ | | Nirmalakhandan et al. (1997) | Q | |
| | $5.4\times10^{-6}$ | | Yaws and Yang (1992) | ? | 21 |
| decahydronaphthalene $C_{10}H_{18}$ (decalin) [91-17-8] NNBZCPXTIHJBJL-UHFFFAOYSA-N | $7.2\times10^{-5}$ | 4100 | Ashworth et al. (1988) | M | 278 |
| | $2.1\times10^{-5}$ | | Duchowicz et al. (2020) | V | 186 |
| | $2.1\times10^{-5}$ | | HSDB (2015) | V | |
| | $4.4\times10^{-3}$ | | Duchowicz et al. (2020) | Q | |
| | $6.5\times10^{-5}$ | | Hilal et al. (2008) | Q | |
| | | 4500 | Kühne et al. (2005) | Q | |
| | $2.1\times10^{-5}$ | | Yaffe et al. (2003) | Q | 248, 249 |
| | $6.5\times10^{-5}$ | | Katritzky et al. (1998) | Q | |
| | | 4100 | Kühne et al. (2005) | ? | |
| ($Z$)-bicyclo[4.4.0]decane $C_{10}H_{18}$ (*cis*-decahydronaphthalene; *cis*-decalin) [493-01-6] NNBZCPXTIHJBJL-AOOOYVTPSA-N | $4.3\times10^{-4}$ | | Mackay et al. (1993) | V | |
| | $6.1\times10^{-5}$ | | Yaws (2003) | X | 237 |
| | $4.3\times10^{-5}$ | | Gharagheizi et al. (2012) | Q | |
| | $8.1\times10^{-5}$ | | Gharagheizi et al. (2010) | Q | 246 |
| | $6.1\times10^{-5}$ | | Yaws (1999) | ? | 21 |
| ($E$)-bicyclo[4.4.0]decane $C_{10}H_{18}$ (*trans*-decahydronaphthalene; *trans*-decalin) [493-02-7] NNBZCPXTIHJBJL-MGCOHNPYSA-N | $2.7\times10^{-4}$ | | Mackay et al. (1993) | V | |
| | $3.9\times10^{-5}$ | | Yaws (2003) | X | 237 |
| | $2.6\times10^{-5}$ | | Gharagheizi et al. (2012) | Q | |
| | $8.1\times10^{-5}$ | | Gharagheizi et al. (2010) | Q | 246 |
| | $4.0\times10^{-5}$ | | Yaws (1999) | ? | 21 |
| 2,6,6-trimethylbicyclo[3.1.1]heptane $C_{10}H_{18}$ (dihydropinene) [473-55-2] XOKSLPVRUOBDEW-UHFFFAOYSA-N | $2.8\times10^{-5}$ | | HSDB (2015) | Q | 99 |
| tricyclene $C_{10}H_{16}$ [508-32-7] RRBYUSWBLVXTQN-UHFFFAOYSA-N | $4.9\times10^{-5}$ | | Plyasunov and Shock (2000) | L | |
| hexylcyclopentane $C_{11}H_{22}$ [4457-00-5] LKHGKBBAJAFMSQ-UHFFFAOYSA-N | $5.1\times10^{-6}$ | | Yaws (2003) | X | 237 |
| | $6.2\times10^{-6}$ | | Gharagheizi et al. (2012) | Q | |
| | $6.5\times10^{-6}$ | | Gharagheizi et al. (2010) | Q | 246 |
| pentylcyclohexane $C_{11}H_{22}$ [4292-92-6] HLTMUYBTNSVOFY-UHFFFAOYSA-N | $7.1\times10^{-6}$ | | Yaws (2003) | X | 237 |
| | $6.6\times10^{-6}$ | | Gharagheizi et al. (2012) | Q | |
| | $9.2\times10^{-6}$ | | Gharagheizi et al. (2010) | Q | 246 |



Table A2.2: Cycloalkanes (...continued)

| Substance Formula (Trivial Name) [CAS Registry Number] InChIKey | $H_s^{cp}$ (at $T^\ominus$) $\left[\dfrac{\text{mol}}{\text{m}^3\,\text{Pa}}\right]$ | $\dfrac{\mathrm{d}\ln H_s^{cp}}{\mathrm{d}(1/T)}$ [K] | Reference | Type | Note |
|---|---|---|---|---|---|
| cyclododecane $C_{12}H_{24}$ [294-62-2] DDTBPAQBQHZRDW-UHFFFAOYSA-N | $6.4\times10^{-6}$ | | HSDB (2015) | Q | 99 |
| heptylcyclopentane $C_{12}H_{24}$ [5617-42-5] BOFNAOHMSHEKQL-UHFFFAOYSA-N | $6.6\times10^{-6}$ $9.8\times10^{-6}$ $7.3\times10^{-6}$ | | Yaws (2003) Gharagheizi et al. (2012) Gharagheizi et al. (2010) | X Q Q | 237 246 |
| hexylcyclohexane $C_{12}H_{24}$ [4292-75-5] QHWAQXOSHHKCFK-UHFFFAOYSA-N | $8.8\times10^{-6}$ $7.4\times10^{-6}$ $1.1\times10^{-5}$ | | Yaws (2003) Gharagheizi et al. (2012) Gharagheizi et al. (2010) | X Q Q | 237 246 |
| 1,1'-bicyclohexyl $C_{12}H_{22}$ [92-51-3] WVIIMZNLDWSIRH-UHFFFAOYSA-N | $3.1\times10^{-5}$ | | Hilal et al. (2008) | Q | |
| heptylcyclohexane $C_{13}H_{26}$ [5617-41-4] MSTLSCNJAHAQNU-UHFFFAOYSA-N | $1.4\times10^{-5}$ $1.3\times10^{-5}$ $1.7\times10^{-5}$ | | Yaws (2003) Gharagheizi et al. (2012) Gharagheizi et al. (2010) | X Q Q | 237 246 |
| octylcyclopentane $C_{13}H_{26}$ [1795-20-6] HARXDULSBROLME-UHFFFAOYSA-N | $1.1\times10^{-5}$ $1.8\times10^{-5}$ $1.0\times10^{-5}$ | | Yaws (2003) Gharagheizi et al. (2012) Gharagheizi et al. (2010) | X Q Q | 237 246 |
| octylcyclohexane $C_{14}H_{28}$ [1795-15-9] FBXWCEKQCVOOLT-UHFFFAOYSA-N | $3.1\times10^{-5}$ $2.1\times10^{-5}$ $3.3\times10^{-5}$ | | Yaws (2003) Gharagheizi et al. (2012) Gharagheizi et al. (2010) | X Q Q | 237 246 |
| nonylcyclopentane $C_{14}H_{28}$ [2882-98-6] GDCYEUOAZVKNHT-UHFFFAOYSA-N | $2.3\times10^{-5}$ $2.0\times10^{-5}$ $1.8\times10^{-5}$ | | Yaws (2003) Gharagheizi et al. (2012) Gharagheizi et al. (2010) | X Q Q | 237 246 |
| octahydro-1,1,2,3,3-pentamethyl-1H-indene $C_{14}H_{26}$ [33704-60-8] TUALLFJCLUYJEN-UHFFFAOYSA-N | $9.0\times10^{-6}$ $1.1\times10^{-6}$ $6.5\times10^{-4}$ $3.5\times10^{-5}$ | | Zhang et al. (2010) Zhang et al. (2010) Zhang et al. (2010) Zhang et al. (2010) | Q Q Q Q | 287, 288 287, 289 287, 290 287, 291 |
| nonylcyclohexane $C_{15}H_{30}$ [2883-02-5] CLMFECCMAVQYQA-UHFFFAOYSA-N | $8.7\times10^{-5}$ $2.2\times10^{-5}$ $8.7\times10^{-5}$ | | Yaws (2003) Gharagheizi et al. (2012) Gharagheizi et al. (2010) | X Q Q | 237 246 |



Table A2.2: Cycloalkanes (. . . continued)

| Substance Formula (Trivial Name) [CAS Registry Number] InChIKey | $H_s^{cp}$ (at $T^\ominus$) $\left[\dfrac{\text{mol}}{\text{m}^3\,\text{Pa}}\right]$ | $\dfrac{\text{d}\ln H_s^{cp}}{\text{d}(1/T)}$ [K] | Reference | Type | Note |
|---|---|---|---|---|---|
| decylcyclopentane | $6.4\times10^{-5}$ | | Yaws (2003) | X | 237 |
| $C_{15}H_{30}$ | $2.0\times10^{-5}$ | | Gharagheizi et al. (2012) | Q | |
| [1795-21-7] | $4.2\times10^{-5}$ | | Gharagheizi et al. (2010) | Q | 246 |
| WOUVLFFZQCUYOL-UHFFFAOYSA-N | | | | | |
| decylcyclohexane | $3.7\times10^{-4}$ | | Yaws (1999) | ? | 21 |
| $C_{16}H_{32}$ | | | | | |
| [1795-16-0] | | | | | |
| STWFZICHPLEOIC-UHFFFAOYSA-N | | | | | |
| 1,1'-(2-methylpentane-2,4-diyl)dicyclohexane | $2.9\times10^{-6}$ | | Zhang et al. (2010) | Q | 287, 288 |
| $C_{18}H_{34}$ | $2.1\times10^{-6}$ | | Zhang et al. (2010) | Q | 287, 289 |
| [38970-72-8] | $1.1\times10^{-3}$ | | Zhang et al. (2010) | Q | 287, 290 |
| XUVKLBIJXLIPDZ-UHFFFAOYSA-N | $1.9\times10^{-5}$ | | Zhang et al. (2010) | Q | 287, 291 |
| 1,1':3',1''-tercyclohexane | $6.7\times10^{-6}$ | | Zhang et al. (2010) | Q | 287, 288 |
| $C_{18}H_{32}$ | $1.5\times10^{-5}$ | | Zhang et al. (2010) | Q | 287, 289 |
| [1706-50-9] | $1.7\times10^{-3}$ | | Zhang et al. (2010) | Q | 287, 290 |
| JBQRJHVXZMSLNH-UHFFFAOYSA-N | $9.0\times10^{-5}$ | | Zhang et al. (2010) | Q | 287, 291 |



### A2.3 Aliphatic alkenes and cycloalkenes

Table A2.3: Aliphatic alkenes and cycloalkenes

| Substance<br>Formula<br>(Trivial Name)<br>[CAS Registry Number]<br>InChIKey | $H_s^{cp}$<br>(at $T^{\ominus}$)<br>$\left[\dfrac{\text{mol}}{\text{m}^3\,\text{Pa}}\right]$ | $\dfrac{\mathrm{d}\ln H_s^{cp}}{\mathrm{d}(1/T)}$<br><br>[K] | Reference | Type | Note |
|---|---|---|---|---|---|
| ethene | $5.9\times10^{-5}$ | 2200 | Burkholder et al. (2019) | L | 1 |
| $C_2H_4$ | $5.9\times10^{-5}$ | 2200 | Burkholder et al. (2015) | L | 1 |
| (ethylene) | $5.9\times10^{-5}$ | 2200 | Sander et al. (2011) | L | 1 |
| [74-85-1] | $5.9\times10^{-5}$ | 2200 | Sander et al. (2006) | L | 1 |
| VGGSQFUCUMXWEO-UHFFFAOYSA-N | $4.8\times10^{-5}$ | 2000 | Plyasunov and Shock (2000) | L | |
| | $4.7\times10^{-5}$ | 2000 | Hayduk (1994) | L | 1 |
| | $4.6\times10^{-5}$ | | Mackay and Shiu (1981) | L | |
| | $4.7\times10^{-5}$ | 1800 | Wilhelm et al. (1977) | L | |
| | $3.5\times10^{-5}$ | | Steward et al. (1973) | L | 14 |
| | $4.6\times10^{-5}$ | 2200 | Allott et al. (1973) | L | |
| | $4.9\times10^{-5}$ | 2000 | Maaßen (1995) | M | 306 |
| | $4.8\times10^{-5}$ | 1900 | Reichl (1995) | M | 307 |
| | $4.7\times10^{-5}$ | | McAuliffe (1966) | M | |
| | $4.7\times10^{-5}$ | 2000 | Morrison and Billett (1952) | M | 308 |
| | $4.8\times10^{-5}$ | | Orcutt and Seevers (1937a) | M | |
| | $3.4\times10^{-5}$ | | Grollman (1929) | M | 58 |
| | $4.8\times10^{-5}$ | 2300 | Winkler (1906) | M | |
| | $4.6\times10^{-5}$ | | Hine and Mookerjee (1975) | V | |
| | $4.7\times10^{-5}$ | 1900 | Wauchope and Haque (1972) | V | |
| | $3.1\times10^{-5}$ | | Pierotti (1965) | T | |
| | $4.7\times10^{-5}$ | | Yaws (2003) | X | 237 |
| | $4.7\times10^{-5}$ | | Deno and Berkheimer (1960) | C | |
| | $3.4\times10^{-5}$ | | Hayer et al. (2022) | Q | 20 |
| | $2.0\times10^{-5}$ | | Keshavarz et al. (2022) | Q | |
| | $4.0\times10^{-3}$ | | Duchowicz et al. (2020) | Q | 299 |
| | $2.1\times10^{-4}$ | | Wang et al. (2017) | Q | 80, 238 |
| | $2.6\times10^{-5}$ | | Wang et al. (2017) | Q | 80, 239 |
| | $8.3\times10^{-5}$ | | Wang et al. (2017) | Q | 80, 240 |
| | $4.6\times10^{-5}$ | | Li et al. (2014) | Q | 241 |
| | $2.4\times10^{-5}$ | | Gharagheizi et al. (2012) | Q | |
| | $7.8\times10^{-5}$ | | Raventos-Duran et al. (2010) | Q | 242, 243 |
| | $2.5\times10^{-5}$ | | Raventos-Duran et al. (2010) | Q | 244 |
| | $9.9\times10^{-5}$ | | Raventos-Duran et al. (2010) | Q | 245 |
| | $6.8\times10^{-5}$ | | Gharagheizi et al. (2010) | Q | 246 |
| | $2.9\times10^{-5}$ | | Hilal et al. (2008) | Q | |
| | $1.0\times10^{-4}$ | | Modarresi et al. (2007) | Q | 67 |
| | | 2700 | Kühne et al. (2005) | Q | |
| | $4.7\times10^{-5}$ | | Yaffe et al. (2003) | Q | 248, 249 |
| | $8.2\times10^{-5}$ | | English and Carroll (2001) | Q | 230, 231 |
| | $1.5\times10^{-5}$ | | Katritzky et al. (1998) | Q | |
| | $9.5\times10^{-5}$ | | Suzuki et al. (1992) | Q | 232 |
| | $5.2\times10^{-5}$ | | Nirmalakhandan and Speece (1988) | Q | |
| | $4.3\times10^{-5}$ | | Duchowicz et al. (2020) | ? | 185, 21 |
| | | 1900 | Kühne et al. (2005) | ? | |
| | $4.8\times10^{-5}$ | | Yaws (1999) | ? | 21 |



Table A2.3: Aliphatic alkenes and cycloalkenes (. . . continued)

| Substance Formula (Trivial Name) [CAS Registry Number] InChIKey | $H_s^{cp}$ (at $T^{\ominus}$) $\left[\dfrac{\mathrm{mol}}{\mathrm{m^3\,Pa}}\right]$ | $\dfrac{\mathrm{d}\ln H_s^{cp}}{\mathrm{d}(1/T)}$ [K] | Reference | Type | Note |
|---|---|---|---|---|---|
| | $5.1\times10^{-5}$ | 2400 | Yaws et al. (1999) | ? | 21 |
| | $3.9\times10^{-5}$ | | Abraham and Weathersby (1994) | ? | 21 |
| | $4.8\times10^{-5}$ | 2000 | Dean and Lange (1999) | ? | 309, 23 |
| | $4.7\times10^{-5}$ | | Yaws and Yang (1992) | ? | 21 |
| | $4.6\times10^{-5}$ | | Abraham et al. (1990) | ? | |
| | $4.8\times10^{-5}$ | | Seinfeld (1986) | ? | 21 |
| propene $C_3H_6$ (propylene) [115-07-1] QQONPFPTGQHPMA-UHFFFAOYSA-N | $5.6\times10^{-5}$ | 2600 | Plyasunov and Shock (2000) | L | |
| | $4.7\times10^{-5}$ | | Mackay and Shiu (1981) | L | |
| | $7.3\times10^{-5}$ | 3400 | Wilhelm et al. (1977) | L | |
| | $5.4\times10^{-5}$ | 2700 | Maaßen (1995) | M | 310 |
| | $5.4\times10^{-5}$ | 2800 | Reichl (1995) | M | 311 |
| | $4.8\times10^{-5}$ | | McAuliffe (1966) | M | |
| | $4.7\times10^{-5}$ | | Hine and Mookerjee (1975) | V | |
| | $4.4\times10^{-5}$ | | Irmann (1965) | V | |
| | $4.8\times10^{-5}$ | | Yaws (2003) | X | 237 |
| | $9.2\times10^{-5}$ | | Deno and Berkheimer (1960) | C | |
| | $6.5\times10^{-5}$ | | Hayer et al. (2022) | Q | 20 |
| | $2.7\times10^{-5}$ | | Keshavarz et al. (2022) | Q | |
| | $1.6\times10^{-3}$ | | Duchowicz et al. (2020) | Q | 299 |
| | $2.5\times10^{-4}$ | | Wang et al. (2017) | Q | 80, 238 |
| | $2.7\times10^{-5}$ | | Wang et al. (2017) | Q | 80, 239 |
| | $8.5\times10^{-5}$ | | Wang et al. (2017) | Q | 80, 240 |
| | $2.9\times10^{-5}$ | | Gharagheizi et al. (2012) | Q | |
| | $6.2\times10^{-5}$ | | Raventos-Duran et al. (2010) | Q | 242, 243 |
| | $9.9\times10^{-5}$ | | Raventos-Duran et al. (2010) | Q | 244 |
| | $6.2\times10^{-5}$ | | Raventos-Duran et al. (2010) | Q | 245 |
| | $6.6\times10^{-5}$ | | Gharagheizi et al. (2010) | Q | 246 |
| | $3.4\times10^{-5}$ | | Hilal et al. (2008) | Q | |
| | $5.9\times10^{-5}$ | | Modarresi et al. (2007) | Q | 67 |
| | | 3100 | Kühne et al. (2005) | Q | |
| | $5.1\times10^{-5}$ | | Yaffe et al. (2003) | Q | 248, 249 |
| | $1.0\times10^{-4}$ | | Yao et al. (2002) | Q | 229 |
| | $6.2\times10^{-5}$ | | English and Carroll (2001) | Q | 230, 231 |
| | $4.4\times10^{-5}$ | | Katritzky et al. (1998) | Q | |
| | $6.9\times10^{-5}$ | | Suzuki et al. (1992) | Q | 232 |
| | $4.1\times10^{-5}$ | | Nirmalakhandan and Speece (1988) | Q | |
| | $4.6\times10^{-5}$ | | Irmann (1965) | Q | |
| | $5.0\times10^{-5}$ | | Duchowicz et al. (2020) | ? | 185, 21 |
| | | 3800 | Kühne et al. (2005) | ? | |
| | $4.8\times10^{-5}$ | | Yaws (1999) | ? | 21 |
| | $6.8\times10^{-5}$ | 2800 | Yaws et al. (1999) | ? | 21 |
| | $4.8\times10^{-5}$ | | Yaws and Yang (1992) | ? | 21 |
| | $4.3\times10^{-5}$ | | Abraham et al. (1990) | ? | |



Table A2.3: Aliphatic alkenes and cycloalkenes (...continued)

| Substance Formula (Trivial Name) [CAS Registry Number] InChIKey | $H_s^{cp}$ (at $T^{\ominus}$) $\left[\dfrac{\mathrm{mol}}{\mathrm{m^3\,Pa}}\right]$ | $\dfrac{\mathrm{d}\ln H_s^{cp}}{\mathrm{d}(1/T)}$ [K] | Reference | Type | Note |
|---|---|---|---|---|---|
| 1-butene | $4.2\times10^{-5}$ | 3000 | Plyasunov and Shock (2000) | L | |
| $C_4H_8$ | $1.3\times10^{-4}$ | 6400 | Wilhelm et al. (1977) | L | |
| [106-98-9] | $4.5\times10^{-5}$ | 3000 | Serra and Palavra (2003) | M | 312 |
| VXNZUUAINFGPBY-UHFFFAOYSA-N | $4.0\times10^{-5}$ | | McAuliffe (1966) | M | |
| | $3.9\times10^{-5}$ | | Mackay et al. (2006a) | V | |
| | $3.9\times10^{-5}$ | | Mackay et al. (1993) | V | |
| | $3.9\times10^{-5}$ | | Hine and Mookerjee (1975) | V | |
| | $4.1\times10^{-5}$ | | Irmann (1965) | V | |
| | $4.0\times10^{-5}$ | | Yaws (2003) | X | 237 |
| | $5.5\times10^{-5}$ | | Hayer et al. (2022) | Q | 20 |
| | $3.6\times10^{-5}$ | | Keshavarz et al. (2022) | Q | |
| | $1.6\times10^{-3}$ | | Duchowicz et al. (2020) | Q | 184 |
| | $2.2\times10^{-4}$ | | Wang et al. (2017) | Q | 80, 238 |
| | $3.2\times10^{-5}$ | | Wang et al. (2017) | Q | 80, 239 |
| | $8.5\times10^{-5}$ | | Wang et al. (2017) | Q | 80, 240 |
| | $3.9\times10^{-5}$ | | Li et al. (2014) | Q | 241 |
| | $3.1\times10^{-5}$ | | Gharagheizi et al. (2012) | Q | |
| | $4.9\times10^{-5}$ | | Raventos-Duran et al. (2010) | Q | 242, 243 |
| | $6.2\times10^{-5}$ | | Raventos-Duran et al. (2010) | Q | 244 |
| | $4.9\times10^{-5}$ | | Raventos-Duran et al. (2010) | Q | 245 |
| | $4.6\times10^{-5}$ | | Gharagheizi et al. (2010) | Q | 246 |
| | $3.4\times10^{-5}$ | | Hilal et al. (2008) | Q | |
| | $5.7\times10^{-5}$ | | Modarresi et al. (2007) | Q | 67 |
| | $6.0\times10^{-6}$ | | Modarresi et al. (2005) | Q | 247 |
| | $3.9\times10^{-5}$ | | Yaffe et al. (2003) | Q | 248, 249 |
| | $7.1\times10^{-5}$ | | Yao et al. (2002) | Q | 229 |
| | $4.7\times10^{-5}$ | | English and Carroll (2001) | Q | 230, 231 |
| | $5.6\times10^{-5}$ | | Katritzky et al. (1998) | Q | |
| | $5.2\times10^{-5}$ | | Suzuki et al. (1992) | Q | 232 |
| | $3.4\times10^{-5}$ | | Nirmalakhandan and Speece (1988) | Q | |
| | $4.1\times10^{-5}$ | | Irmann (1965) | Q | |
| | $4.2\times10^{-5}$ | | Duchowicz et al. (2020) | ? | 185, 21 |
| | $4.0\times10^{-5}$ | | Yaws (1999) | ? | 21 |
| | $4.0\times10^{-5}$ | | Yaws and Yang (1992) | ? | 21 |
| | $3.9\times10^{-5}$ | | Abraham et al. (1990) | ? | |
| | | | Mackay and Shiu (1981) | W | 313 |
| 2-butene | $5.1\times10^{-5}$ | | Hilal et al. (2008) | Q | |
| $C_4H_8$ | | | | | |
| [107-01-7] | | | | | |
| IAQRGUVFOMOMEM-UHFFFAOYSA-N | | | | | |





Table A2.3: Aliphatic alkenes and cycloalkenes (...continued)

| Substance Formula (Trivial Name) [CAS Registry Number] InChIKey | $H_s^{cp}$ (at $T^\ominus$) $\left[\dfrac{\text{mol}}{\text{m}^3\,\text{Pa}}\right]$ | $\dfrac{\text{d}\ln H_s^{cp}}{\text{d}(1/T)}$ [K] | Reference | Type | Note |
|---|---|---|---|---|---|
| *cis*-2-butene | $5.5\times10^{-5}$ | | Irmann (1965) | V | |
| C$_4$H$_8$ | $4.1\times10^{-5}$ | | Yaws (2003) | X | 237 |
| [590-18-1] | $3.6\times10^{-5}$ | | Keshavarz et al. (2022) | Q | |
| IAQRGUVFOMOMEM-ARJAWSKDSA-N | $5.2\times10^{-4}$ | | Duchowicz et al. (2020) | Q | 299 |
| | $3.2\times10^{-4}$ | | Wang et al. (2017) | Q | 314, 80, 238 |
| | $5.4\times10^{-5}$ | | Wang et al. (2017) | Q | 314, 80, 239 |
| | $6.0\times10^{-5}$ | | Wang et al. (2017) | Q | 314, 80, 240 |
| | $3.2\times10^{-5}$ | | Gharagheizi et al. (2012) | Q | |
| | $5.7\times10^{-5}$ | | Gharagheizi et al. (2010) | Q | 246 |
| | $4.5\times10^{-5}$ | | Modarresi et al. (2007) | Q | 67 |
| | $6.9\times10^{-6}$ | | Modarresi et al. (2005) | Q | 247 |
| | $9.1\times10^{-5}$ | | Yao et al. (2002) | Q | 229 |
| | $5.9\times10^{-5}$ | | Irmann (1965) | Q | |
| | $4.3\times10^{-5}$ | | Duchowicz et al. (2020) | ? | 185, 21 |
| | $4.1\times10^{-5}$ | | Yaws (1999) | ? | 21 |
| *trans*-2-butene | $3.9\times10^{-5}$ | | Irmann (1965) | V | |
| C$_4$H$_8$ | $4.3\times10^{-5}$ | | Yaws (2003) | X | 237 |
| [624-64-6] | $1.0\times10^{-4}$ | | Hayer et al. (2022) | Q | 20 |
| IAQRGUVFOMOMEM-ONEGZZNKSA-N | $3.6\times10^{-5}$ | | Keshavarz et al. (2022) | Q | |
| | $5.2\times10^{-4}$ | | Duchowicz et al. (2020) | Q | |
| | $3.2\times10^{-4}$ | | Wang et al. (2017) | Q | 314, 80, 238 |
| | $5.4\times10^{-5}$ | | Wang et al. (2017) | Q | 314, 80, 239 |
| | $6.0\times10^{-5}$ | | Wang et al. (2017) | Q | 314, 80, 240 |
| | $2.7\times10^{-5}$ | | Gharagheizi et al. (2012) | Q | |
| | $5.7\times10^{-5}$ | | Gharagheizi et al. (2010) | Q | 246 |
| | $4.5\times10^{-5}$ | | Modarresi et al. (2007) | Q | 67 |
| | $8.5\times10^{-6}$ | | Modarresi et al. (2005) | Q | 247 |
| | $7.9\times10^{-5}$ | | Yao et al. (2002) | Q | 229 |
| | $5.4\times10^{-5}$ | | Irmann (1965) | Q | |
| | $4.4\times10^{-5}$ | | Duchowicz et al. (2020) | ? | 185, 21 |
| | $4.3\times10^{-5}$ | | Yaws (1999) | ? | 21 |
| 2-methylpropene | $5.7\times10^{-5}$ | 3000 | Plyasunov and Shock (2000) | L | |
| C$_4$H$_8$ | $5.6\times10^{-5}$ | 3000 | Wilhelm et al. (1977) | L | |
| (isobutene) | $4.8\times10^{-5}$ | | McAuliffe (1966) | M | |
| [115-11-7] | $4.6\times10^{-5}$ | | Mackay et al. (2006a) | V | |
| VQTUBCCKSQIDNK-UHFFFAOYSA-N | $4.6\times10^{-5}$ | | Mackay et al. (1993) | V | |
| | $4.6\times10^{-5}$ | | Hine and Mookerjee (1975) | V | |
| | $4.8\times10^{-5}$ | | Yaws (2003) | X | 237 |
| | $6.1\times10^{-5}$ | | Hayer et al. (2022) | Q | 20 |
| | $3.6\times10^{-5}$ | | Keshavarz et al. (2022) | Q | |



Table A2.3: Aliphatic alkenes and cycloalkenes (…continued)

| Substance<br>Formula<br>(Trivial Name)<br>[CAS Registry Number]<br>InChIKey | $H_s^{cp}$<br>(at $T^\ominus$)<br>$\left[\dfrac{\mathrm{mol}}{\mathrm{m^3\,Pa}}\right]$ | $\dfrac{\mathrm{d}\ln H_s^{cp}}{\mathrm{d}(1/T)}$<br><br>[K] | Reference | Type | Note |
|---|---|---|---|---|---|
| | $5.1\times10^{-4}$ | | Duchowicz et al. (2020) | Q | 299 |
| | $1.6\times10^{-4}$ | | Wang et al. (2017) | Q | 80, 238 |
| | $3.9\times10^{-5}$ | | Wang et al. (2017) | Q | 80, 239 |
| | $9.3\times10^{-5}$ | | Wang et al. (2017) | Q | 80, 240 |
| | $2.0\times10^{-5}$ | | Gharagheizi et al. (2012) | Q | |
| | $4.9\times10^{-5}$ | | Raventos-Duran et al. (2010) | Q | 242, 243 |
| | $9.9\times10^{-5}$ | | Raventos-Duran et al. (2010) | Q | 244 |
| | $3.9\times10^{-5}$ | | Raventos-Duran et al. (2010) | Q | 245 |
| | $6.1\times10^{-5}$ | | Gharagheizi et al. (2010) | Q | 246 |
| | $8.6\times10^{-5}$ | | Hilal et al. (2008) | Q | |
| | $4.6\times10^{-5}$ | | Modarresi et al. (2007) | Q | 67 |
| | | 3400 | Kühne et al. (2005) | Q | |
| | $8.2\times10^{-6}$ | | Modarresi et al. (2005) | Q | 247 |
| | $4.7\times10^{-5}$ | | Yaffe et al. (2003) | Q | 248, 249 |
| | $6.5\times10^{-5}$ | | Yao et al. (2002) | Q | 229 |
| | $4.8\times10^{-5}$ | | English and Carroll (2001) | Q | 230, 231 |
| | $5.3\times10^{-5}$ | | Katritzky et al. (1998) | Q | |
| | $4.5\times10^{-5}$ | | Suzuki et al. (1992) | Q | 232 |
| | $2.8\times10^{-5}$ | | Nirmalakhandan and Speece (1988) | Q | |
| | $4.5\times10^{-5}$ | | Duchowicz et al. (2020) | ? | 185, 21 |
| | | 3000 | Kühne et al. (2005) | ? | |
| | $4.8\times10^{-5}$ | | Yaws (1999) | ? | 21 |
| | $5.6\times10^{-5}$ | 3000 | Yaws et al. (1999) | ? | 21 |
| | $4.8\times10^{-5}$ | | Yaws and Yang (1992) | ? | 21 |
| | | | Mackay and Shiu (1981) | W | 313 |
| 1-pentene<br>$C_5H_{10}$<br>[109-67-1]<br>YWAKXRMUMFPDSH-UHFFFAOYSA-N | $3.0\times10^{-5}$ | | Brockbank (2013) | L | |
| | $2.4\times10^{-5}$ | | Plyasunov and Shock (2000) | L | |
| | $2.5\times10^{-5}$ | | Mackay and Shiu (1981) | L | |
| | $2.5\times10^{-5}$ | | Duchowicz et al. (2020) | V | 186 |
| | $2.5\times10^{-5}$ | | HSDB (2015) | V | |
| | $2.5\times10^{-5}$ | | Mackay et al. (2006a) | V | |
| | $2.5\times10^{-5}$ | | Mackay et al. (1993) | V | |
| | $2.5\times10^{-5}$ | | Eastcott et al. (1988) | V | |
| | $1.8\times10^{-5}$ | | Amoore and Buttery (1978) | V | |
| | $2.4\times10^{-5}$ | | Hine and Mookerjee (1975) | V | |
| | $2.7\times10^{-5}$ | | McAuliffe (1966) | V | 24 |
| | $2.5\times10^{-5}$ | | Yaws (2003) | X | 237 |
| | $1.6\times10^{-3}$ | | Duchowicz et al. (2020) | Q | |
| | $1.7\times10^{-4}$ | | Wang et al. (2017) | Q | 80, 238 |
| | $2.3\times10^{-5}$ | | Wang et al. (2017) | Q | 80, 239 |
| | $7.6\times10^{-5}$ | | Wang et al. (2017) | Q | 80, 240 |
| | $3.0\times10^{-5}$ | | Gharagheizi et al. (2012) | Q | |
| | $3.9\times10^{-5}$ | | Raventos-Duran et al. (2010) | Q | 271, 243 |
| | $3.1\times10^{-5}$ | | Raventos-Duran et al. (2010) | Q | 244 |
| | $3.9\times10^{-5}$ | | Raventos-Duran et al. (2010) | Q | 245 |
| | $3.2\times10^{-5}$ | | Gharagheizi et al. (2010) | Q | 246 |
| | $2.5\times10^{-5}$ | | Hilal et al. (2008) | Q | |



Table A2.3: Aliphatic alkenes and cycloalkenes (...continued)

| Substance Formula (Trivial Name) [CAS Registry Number] InChIKey | $H_s^{cp}$ (at $T^\ominus$) $\left[\dfrac{\text{mol}}{\text{m}^3\,\text{Pa}}\right]$ | $\dfrac{\text{d}\ln H_s^{cp}}{\text{d}(1/T)}$ [K] | Reference | Type | Note |
|---|---|---|---|---|---|
| | $4.7\times10^{-5}$ | | Modarresi et al. (2007) | Q | 67 |
| | $5.4\times10^{-6}$ | | Modarresi et al. (2005) | Q | 247 |
| | $2.5\times10^{-5}$ | | Yaffe et al. (2003) | Q | 248, 249 |
| | $5.1\times10^{-5}$ | | Yao et al. (2002) | Q | 229 |
| | $3.6\times10^{-5}$ | | English and Carroll (2001) | Q | 230, 260 |
| | $6.0\times10^{-5}$ | | Katritzky et al. (1998) | Q | |
| | $4.0\times10^{-5}$ | | Suzuki et al. (1992) | Q | 232 |
| | $2.7\times10^{-5}$ | | Nirmalakhandan and Speece (1988) | Q | |
| | $2.5\times10^{-5}$ | | Yaws (1999) | ? | 21 |
| | $2.5\times10^{-5}$ | | Yaws and Yang (1992) | ? | 21 |
| | $2.4\times10^{-5}$ | | Abraham et al. (1990) | ? | |
| 2-pentene C$_5$H$_{10}$ [109-68-2] QMMOXUPEWRXHJS-UHFFFAOYSA-N | $4.1\times10^{-5}$ | | Plyasunov and Shock (2000) | L | |
| | $4.1\times10^{-5}$ | | Duchowicz et al. (2020) | V | 186 |
| | $4.4\times10^{-5}$ | | Eastcott et al. (1988) | V | |
| | $5.2\times10^{-4}$ | | Duchowicz et al. (2020) | Q | |
| | $3.6\times10^{-5}$ | | Hilal et al. (2008) | Q | |
| | $4.7\times10^{-5}$ | | Yaffe et al. (2003) | Q | 248, 272 |
| | $6.1\times10^{-5}$ | | Katritzky et al. (1998) | Q | |
| *cis*-2-pentene C$_5$H$_{10}$ [627-20-3] QMMOXUPEWRXHJS-HYXAFXHYSA-N | $4.4\times10^{-5}$ | | Brockbank (2013) | L | |
| | $4.4\times10^{-5}$ | | Mackay and Shiu (1981) | L | |
| | $4.4\times10^{-5}$ | | Duchowicz et al. (2020) | V | 186 |
| | $4.4\times10^{-5}$ | | Yaws (2003) | X | 237 |
| | $5.2\times10^{-4}$ | | Duchowicz et al. (2020) | Q | |
| | $2.6\times10^{-4}$ | | Wang et al. (2017) | Q | 314, 80, 238 |
| | $3.7\times10^{-5}$ | | Wang et al. (2017) | Q | 314, 80, 239 |
| | $5.8\times10^{-5}$ | | Wang et al. (2017) | Q | 314, 80, 240 |
| | $4.5\times10^{-5}$ | | HSDB (2015) | Q | 99 |
| | $2.7\times10^{-5}$ | | Gharagheizi et al. (2012) | Q | |
| | $3.6\times10^{-5}$ | | Gharagheizi et al. (2010) | Q | 246 |
| | $5.4\times10^{-6}$ | | Modarresi et al. (2005) | Q | 247 |
| | $4.7\times10^{-5}$ | | Yaffe et al. (2003) | Q | 248, 249 |
| | $7.3\times10^{-5}$ | | Yao et al. (2002) | Q | 229 |
| | $4.4\times10^{-5}$ | | Yaws (1999) | ? | 21 |
| | $4.4\times10^{-5}$ | | Yaws and Yang (1992) | ? | 21 |
| *trans*-2-pentene C$_5$H$_{10}$ [646-04-8] QMMOXUPEWRXHJS-HWKANZROSA-N | $4.3\times10^{-5}$ | | Brockbank (2013) | L | |
| | $4.3\times10^{-5}$ | | Duchowicz et al. (2020) | V | 186 |
| | $4.2\times10^{-5}$ | | Hine and Mookerjee (1975) | V | |
| | $4.3\times10^{-5}$ | | Yaws (2003) | X | 237 |
| | $5.2\times10^{-4}$ | | Duchowicz et al. (2020) | Q | |
| | $2.6\times10^{-4}$ | | Wang et al. (2017) | Q | 314, 80, 238 |
| | $3.7\times10^{-5}$ | | Wang et al. (2017) | Q | 314, 80, 239 |





Table A2.3: Aliphatic alkenes and cycloalkenes (...continued)

| Substance Formula (Trivial Name) [CAS Registry Number] InChIKey | $H_s^{cp}$ (at $T^{\ominus}$) $\left[\dfrac{\text{mol}}{\text{m}^3\,\text{Pa}}\right]$ | $\dfrac{\text{d}\ln H_s^{cp}}{\text{d}(1/T)}$ [K] | Reference | Type | Note |
|---|---|---|---|---|---|
| | $5.9\times10^{-5}$ | | Wang et al. (2017) | Q | 314, 80, 240 |
| | $3.1\times10^{-5}$ | | HSDB (2015) | Q | 99 |
| | $2.6\times10^{-5}$ | | Gharagheizi et al. (2012) | Q | |
| | $3.6\times10^{-5}$ | | Gharagheizi et al. (2010) | Q | 246 |
| | $1.0\times10^{-5}$ | | Modarresi et al. (2005) | Q | 247 |
| | $6.7\times10^{-5}$ | | Yao et al. (2002) | Q | 229 |
| | $3.6\times10^{-5}$ | | English and Carroll (2001) | Q | 230, 231 |
| | $4.0\times10^{-5}$ | | Suzuki et al. (1992) | Q | 232 |
| | $2.7\times10^{-5}$ | | Nirmalakhandan and Speece (1988) | Q | |
| | $4.3\times10^{-5}$ | | Yaws (1999) | ? | 21 |
| | $4.3\times10^{-5}$ | | Yaws and Yang (1992) | ? | 21 |
| 2-methyl-1-butene $C_5H_{10}$ [563-46-2] MHNNAWXXUZQSNM-UHFFFAOYSA-N | $2.3\times10^{-5}$ | | Duchowicz et al. (2020) | V | 186 |
| | $2.3\times10^{-5}$ | | HSDB (2015) | V | |
| | $3.3\times10^{-5}$ | | Yaws (2003) | X | 237 |
| | $5.1\times10^{-4}$ | | Duchowicz et al. (2020) | Q | |
| | $1.5\times10^{-4}$ | | Wang et al. (2017) | Q | 80, 238 |
| | $4.6\times10^{-5}$ | | Wang et al. (2017) | Q | 80, 239 |
| | $8.7\times10^{-5}$ | | Wang et al. (2017) | Q | 80, 240 |
| | $1.9\times10^{-5}$ | | Gharagheizi et al. (2012) | Q | |
| | $3.8\times10^{-5}$ | | Gharagheizi et al. (2010) | Q | 246 |
| | $4.2\times10^{-6}$ | | Modarresi et al. (2005) | Q | 247 |
| | $5.1\times10^{-5}$ | | Yao et al. (2002) | Q | 229, 267 |
| | $3.4\times10^{-5}$ | | Yaws (1999) | ? | 21 |
| 2-methyl-2-butene $C_5H_{10}$ [513-35-9] BKOOMYPCSUNDGP-UHFFFAOYSA-N | $5.7\times10^{-5}$ | 3500 | Brockbank (2013) | L | 1 |
| | $6.7\times10^{-5}$ | 3600 | Plyasunov and Shock (2000) | L | |
| | $4.4\times10^{-5}$ | | Duchowicz et al. (2020) | V | 186 |
| | $7.4\times10^{-5}$ | | Mackay et al. (2006a) | V | |
| | $4.4\times10^{-5}$ | | Hine and Mookerjee (1975) | V | |
| | $3.4\times10^{-5}$ | | Yaws (2003) | X | 237 |
| | $1.7\times10^{-4}$ | | Duchowicz et al. (2020) | Q | |
| | $2.1\times10^{-4}$ | | Wang et al. (2017) | Q | 80, 238 |
| | $6.0\times10^{-5}$ | | Wang et al. (2017) | Q | 80, 239 |
| | $6.3\times10^{-5}$ | | Wang et al. (2017) | Q | 80, 240 |
| | $1.6\times10^{-5}$ | | Gharagheizi et al. (2012) | Q | |
| | $4.3\times10^{-5}$ | | Gharagheizi et al. (2010) | Q | 246 |
| | $7.5\times10^{-5}$ | | Hilal et al. (2008) | Q | |
| | $9.7\times10^{-6}$ | | Modarresi et al. (2005) | Q | 247 |
| | $4.7\times10^{-5}$ | | Yaffe et al. (2003) | Q | 248, 249 |
| | $5.4\times10^{-5}$ | | Yao et al. (2002) | Q | 229 |
| | $4.8\times10^{-5}$ | | English and Carroll (2001) | Q | 230, 274 |
| | $5.8\times10^{-5}$ | | Katritzky et al. (1998) | Q | |
| | $3.6\times10^{-5}$ | | Suzuki et al. (1992) | Q | 232 |
| | $2.3\times10^{-5}$ | | Nirmalakhandan and Speece (1988) | Q | |
| | $3.5\times10^{-5}$ | | Yaws (1999) | ? | 21 |



Table A2.3: Aliphatic alkenes and cycloalkenes (. . . continued)

| Substance Formula (Trivial Name) [CAS Registry Number] InChIKey | $H_s^{cp}$ (at $T^{\ominus}$) $\left[\dfrac{\text{mol}}{\text{m}^3\,\text{Pa}}\right]$ | $\dfrac{\text{d}\ln H_s^{cp}}{\text{d}(1/T)}$ [K] | Reference | Type | Note |
|---|---|---|---|---|---|
| 3-methyl-1-butene | $1.9\times10^{-5}$ | | Plyasunov and Shock (2000) | L | |
| $C_5H_{10}$ | $1.8\times10^{-5}$ | | Mackay and Shiu (1981) | L | |
| [563-45-1] | $1.9\times10^{-5}$ | | McAuliffe (1966) | M | |
| YHQXBTXEYZIYOV-UHFFFAOYSA-N | $1.8\times10^{-5}$ | | HSDB (2015) | V | |
| | $1.8\times10^{-5}$ | | Mackay et al. (2006a) | V | |
| | $1.8\times10^{-5}$ | | Mackay et al. (1993) | V | |
| | $1.8\times10^{-5}$ | | Hine and Mookerjee (1975) | V | |
| | $1.9\times10^{-5}$ | | Yaws (2003) | X | 237 |
| | $2.1\times10^{-4}$ | | Wang et al. (2017) | Q | 80, 238 |
| | $1.4\times10^{-5}$ | | Wang et al. (2017) | Q | 80, 239 |
| | $6.9\times10^{-5}$ | | Wang et al. (2017) | Q | 80, 240 |
| | $2.2\times10^{-5}$ | | Gharagheizi et al. (2012) | Q | |
| | $3.9\times10^{-5}$ | | Raventos-Duran et al. (2010) | Q | 242, 243 |
| | $2.0\times10^{-5}$ | | Raventos-Duran et al. (2010) | Q | 244 |
| | $3.9\times10^{-5}$ | | Raventos-Duran et al. (2010) | Q | 245 |
| | $3.7\times10^{-5}$ | | Gharagheizi et al. (2010) | Q | 246 |
| | $1.5\times10^{-5}$ | | Hilal et al. (2008) | Q | |
| | $3.7\times10^{-5}$ | | Modarresi et al. (2007) | Q | 67 |
| | $7.2\times10^{-6}$ | | Modarresi et al. (2005) | Q | 247 |
| | $1.8\times10^{-5}$ | | Yaffe et al. (2003) | Q | 248, 249 |
| | $2.6\times10^{-5}$ | | Yao et al. (2002) | Q | 229 |
| | $3.6\times10^{-5}$ | | English and Carroll (2001) | Q | 230, 231 |
| | $5.7\times10^{-5}$ | | Katritzky et al. (1998) | Q | |
| | $3.6\times10^{-5}$ | | Suzuki et al. (1992) | Q | 232 |
| | $2.3\times10^{-5}$ | | Nirmalakhandan and Speece (1988) | Q | |
| | $1.9\times10^{-5}$ | | Yaws (1999) | ? | 21 |
| | $1.9\times10^{-5}$ | | Yaws and Yang (1992) | ? | 21 |
| 1-hexene | $2.5\times10^{-5}$ | | Brockbank (2013) | L | |
| $C_6H_{12}$ | $2.3\times10^{-5}$ | 3900 | Plyasunov and Shock (2000) | L | |
| [592-41-6] | $2.4\times10^{-5}$ | | Mackay and Shiu (1981) | L | |
| LIKMAJRDDDTEIG-UHFFFAOYSA-N | $2.4\times10^{-5}$ | | Duchowicz et al. (2020) | V | 186 |
| | $2.4\times10^{-5}$ | | HSDB (2015) | V | |
| | $2.4\times10^{-5}$ | | Mackay et al. (2006a) | V | |
| | $2.4\times10^{-5}$ | | Mackay et al. (1993) | V | |
| | $2.4\times10^{-5}$ | | Hwang et al. (1992) | V | |
| | $2.4\times10^{-5}$ | | Eastcott et al. (1988) | V | |
| | $2.4\times10^{-5}$ | | Hine and Mookerjee (1975) | V | |
| | $2.7\times10^{-5}$ | | McAuliffe (1966) | V | 24 |
| | $2.4\times10^{-5}$ | | Yaws (2003) | X | 237 |
| | $1.6\times10^{-3}$ | | Duchowicz et al. (2020) | Q | |
| | $1.4\times10^{-4}$ | | Wang et al. (2017) | Q | 80, 238 |
| | $1.7\times10^{-5}$ | | Wang et al. (2017) | Q | 80, 239 |
| | $7.1\times10^{-5}$ | | Wang et al. (2017) | Q | 80, 240 |
| | $2.8\times10^{-5}$ | | Gharagheizi et al. (2012) | Q | |
| | $2.5\times10^{-5}$ | | Raventos-Duran et al. (2010) | Q | 242, 243 |
| | $2.0\times10^{-5}$ | | Raventos-Duran et al. (2010) | Q | 244 |
| | $2.5\times10^{-5}$ | | Raventos-Duran et al. (2010) | Q | 245 |



Table A2.3: Aliphatic alkenes and cycloalkenes (. . . continued)

| Substance<br>Formula<br>(Trivial Name)<br>[CAS Registry Number]<br>InChIKey | $H_s^{cp}$<br>(at $T^{\ominus}$)<br>$\left[\dfrac{\mathrm{mol}}{\mathrm{m^3\,Pa}}\right]$ | $\dfrac{\mathrm{d}\ln H_s^{cp}}{\mathrm{d}(1/T)}$<br><br>[K] | Reference | Type | Note |
|---|---|---|---|---|---|
| | $2.2\times10^{-5}$ | | Gharagheizi et al. (2010) | Q | 246 |
| | $1.8\times10^{-5}$ | | Hilal et al. (2008) | Q | |
| | $3.6\times10^{-5}$ | | Modarresi et al. (2007) | Q | 67 |
| | | 4100 | Kühne et al. (2005) | Q | |
| | $4.0\times10^{-6}$ | | Modarresi et al. (2005) | Q | 247 |
| | $2.5\times10^{-5}$ | | Yaffe et al. (2003) | Q | 248, 249 |
| | $2.5\times10^{-5}$ | | Yao et al. (2002) | Q | 229, 267 |
| | $2.7\times10^{-5}$ | | English and Carroll (2001) | Q | 230, 231 |
| | $6.1\times10^{-5}$ | | Katritzky et al. (1998) | Q | |
| | $3.1\times10^{-5}$ | | Suzuki et al. (1992) | Q | 232 |
| | $2.1\times10^{-5}$ | | Nirmalakhandan and Speece (1988) | Q | |
| | | 4000 | Kühne et al. (2005) | ? | |
| | $3.3\times10^{-5}$ | | Yaws (1999) | ? | 21 |
| | $3.3\times10^{-5}$ | | Yaws and Yang (1992) | ? | 21 |
| | $2.8\times10^{-5}$ | | Abraham et al. (1990) | ? | |
| *cis*-2-hexene<br>$C_6H_{12}$<br><br>[7688-21-3]<br><br>RYPKRALMXUUNKS-HYXAFXHYSA-N | $2.6\times10^{-5}$ | | Yaws (2003) | X | 237 |
| | $2.3\times10^{-4}$ | | Wang et al. (2017) | Q | 314, 80, 238 |
| | $2.4\times10^{-5}$ | | Wang et al. (2017) | Q | 314, 80, 239 |
| | $5.0\times10^{-5}$ | | Wang et al. (2017) | Q | 314, 80, 240 |
| | $2.3\times10^{-5}$ | | Gharagheizi et al. (2012) | Q | |
| | $2.3\times10^{-5}$ | | Gharagheizi et al. (2010) | Q | 246 |
| | $5.8\times10^{-6}$ | | Modarresi et al. (2005) | Q | 247 |
| | $3.0\times10^{-5}$ | | Yao et al. (2002) | Q | 229 |
| | $2.6\times10^{-5}$ | | Yaws (1999) | ? | 21 |
| *trans*-2-hexene<br>$C_6H_{12}$<br><br>[4050-45-7]<br><br>RYPKRALMXUUNKS-HWKANZROSA-N | $2.6\times10^{-5}$ | | Yaws (2003) | X | 237 |
| | $2.3\times10^{-4}$ | | Wang et al. (2017) | Q | 314, 80, 238 |
| | $2.4\times10^{-5}$ | | Wang et al. (2017) | Q | 314, 80, 239 |
| | $5.0\times10^{-5}$ | | Wang et al. (2017) | Q | 314, 80, 240 |
| | $2.2\times10^{-5}$ | | Gharagheizi et al. (2012) | Q | |
| | $2.3\times10^{-5}$ | | Gharagheizi et al. (2010) | Q | 246 |
| | $5.6\times10^{-6}$ | | Modarresi et al. (2005) | Q | 247 |
| | $3.2\times10^{-5}$ | | Yao et al. (2002) | Q | 229 |
| | $2.7\times10^{-5}$ | | Yaws (1999) | ? | 21 |
| *cis*-3-hexene<br>$C_6H_{12}$<br>[7642-09-3]<br>ZQDPJFUHLCOCRG-WAYWQWQTSA-N | $2.6\times10^{-5}$ | | Yaws (2003) | X | 237 |
| | $2.1\times10^{-5}$ | | Gharagheizi et al. (2012) | Q | |
| | $2.3\times10^{-5}$ | | Gharagheizi et al. (2010) | Q | 246 |
| | $4.5\times10^{-6}$ | | Modarresi et al. (2005) | Q | 247 |
| | $3.3\times10^{-5}$ | | Yao et al. (2002) | Q | 229 |
| | $2.7\times10^{-5}$ | | Yaws (1999) | ? | 21 |



Table A2.3: Aliphatic alkenes and cycloalkenes (...continued)

| Substance Formula (Trivial Name) [CAS Registry Number] InChIKey | $H_s^{cp}$ (at $T^\ominus$) $\left[\dfrac{\text{mol}}{\text{m}^3\,\text{Pa}}\right]$ | $\dfrac{\text{d}\ln H_s^{cp}}{\text{d}(1/T)}$ [K] | Reference | Type | Note |
|---|---|---|---|---|---|
| *trans*-3-hexene | $2.6\times10^{-5}$ | | Yaws (2003) | X | 237 |
| $C_6H_{12}$ | $2.2\times10^{-5}$ | | Gharagheizi et al. (2012) | Q | |
| [13269-52-8] | $2.3\times10^{-5}$ | | Gharagheizi et al. (2010) | Q | 246 |
| ZQDPJFUHLCOCRG-AATRIKPKSA-N | $5.4\times10^{-6}$ | | Modarresi et al. (2005) | Q | 247 |
| | $3.1\times10^{-5}$ | | Yao et al. (2002) | Q | 229 |
| | $2.7\times10^{-5}$ | | Yaws (1999) | ? | 21 |
| 2-methyl-1-pentene | $3.3\times10^{-5}$ | | Plyasunov and Shock (2000) | L | |
| $C_6H_{12}$ | $3.6\times10^{-5}$ | | Mackay and Shiu (1981) | L | |
| [763-29-1] | $3.6\times10^{-5}$ | | Duchowicz et al. (2020) | V | 186 |
| WWUVJRULCWHUSA-UHFFFAOYSA-N | $3.6\times10^{-5}$ | | Mackay et al. (2006a) | V | |
| | $3.6\times10^{-5}$ | | Mackay et al. (1993) | V | |
| | $3.6\times10^{-5}$ | | Eastcott et al. (1988) | V | |
| | $3.4\times10^{-5}$ | | Cabani et al. (1981) | V | |
| | $3.5\times10^{-5}$ | | Yaws (2003) | X | 237 |
| | $5.1\times10^{-4}$ | | Duchowicz et al. (2020) | Q | |
| | $1.9\times10^{-5}$ | | Gharagheizi et al. (2012) | Q | |
| | $2.5\times10^{-5}$ | | Raventos-Duran et al. (2010) | Q | 271, 243 |
| | $3.1\times10^{-5}$ | | Raventos-Duran et al. (2010) | Q | 244 |
| | $2.5\times10^{-5}$ | | Raventos-Duran et al. (2010) | Q | 245 |
| | $2.3\times10^{-5}$ | | Gharagheizi et al. (2010) | Q | 246 |
| | $2.2\times10^{-5}$ | | Hilal et al. (2008) | Q | |
| | $5.8\times10^{-6}$ | | Modarresi et al. (2005) | Q | 247 |
| | $3.6\times10^{-5}$ | | Yaffe et al. (2003) | Q | 248, 249 |
| | $4.5\times10^{-5}$ | | Yao et al. (2002) | Q | 229 |
| | $2.1\times10^{-5}$ | | English and Carroll (2001) | Q | 230, 231 |
| | $1.9\times10^{-5}$ | | Nirmalakhandan et al. (1997) | Q | |
| | $3.6\times10^{-5}$ | | Yaws (1999) | ? | 21 |
| | $3.5\times10^{-5}$ | | Yaws and Yang (1992) | ? | 21 |
| 3-methyl-1-pentene | $2.8\times10^{-5}$ | | Yaws (2003) | X | 237 |
| $C_6H_{12}$ | $2.4\times10^{-5}$ | | Gharagheizi et al. (2012) | Q | |
| [760-20-3] | $2.5\times10^{-5}$ | | Gharagheizi et al. (2010) | Q | 246 |
| LDTAOIUHUHHCMU-UHFFFAOYSA-N | $6.2\times10^{-6}$ | | Modarresi et al. (2005) | Q | 247 |
| | $2.1\times10^{-5}$ | | Yao et al. (2002) | Q | 229 |
| | $2.8\times10^{-5}$ | | Yaws (1999) | ? | 21 |
| 4-methyl-1-pentene | $1.5\times10^{-5}$ | | Plyasunov and Shock (2000) | L | |
| $C_6H_{12}$ | $1.6\times10^{-5}$ | | Mackay and Shiu (1981) | L | |
| [691-37-2] | $1.6\times10^{-5}$ | | Duchowicz et al. (2020) | V | 186 |
| WSSSPWUEQFSQQG-UHFFFAOYSA-N | $1.6\times10^{-5}$ | | Mackay et al. (2006a) | V | |
| | $1.6\times10^{-5}$ | | Mackay et al. (1993) | V | |
| | $1.6\times10^{-5}$ | | Eastcott et al. (1988) | V | |
| | $1.6\times10^{-5}$ | | Hine and Mookerjee (1975) | V | |
| | $1.6\times10^{-5}$ | | Yaws (2003) | X | 237 |
| | $6.1\times10^{-4}$ | | Duchowicz et al. (2020) | Q | |
| | $2.3\times10^{-5}$ | | Gharagheizi et al. (2012) | Q | |
| | $2.5\times10^{-5}$ | | Raventos-Duran et al. (2010) | Q | 242, 243 |
| | $1.6\times10^{-5}$ | | Raventos-Duran et al. (2010) | Q | 244 |



Table A2.3: Aliphatic alkenes and cycloalkenes (...continued)

| Substance Formula (Trivial Name) [CAS Registry Number] InChIKey | $H_s^{cp}$ (at $T^\ominus$) $\left[\dfrac{\text{mol}}{\text{m}^3\,\text{Pa}}\right]$ | $\dfrac{\text{d}\ln H_s^{cp}}{\text{d}(1/T)}$ [K] | Reference | Type | Note |
|---|---|---|---|---|---|
| | $2.5\times10^{-5}$ | | Raventos-Duran et al. (2010) | Q | 245 |
| | $2.5\times10^{-5}$ | | Gharagheizi et al. (2010) | Q | 246 |
| | $1.2\times10^{-5}$ | | Hilal et al. (2008) | Q | |
| | $3.4\times10^{-5}$ | | Modarresi et al. (2007) | Q | 67 |
| | $7.3\times10^{-6}$ | | Modarresi et al. (2005) | Q | 247 |
| | $1.6\times10^{-5}$ | | Yaffe et al. (2003) | Q | 248, 249 |
| | $2.7\times10^{-5}$ | | Yao et al. (2002) | Q | 229 |
| | $2.7\times10^{-5}$ | | English and Carroll (2001) | Q | 230, 231 |
| | $5.8\times10^{-5}$ | | Katritzky et al. (1998) | Q | |
| | $2.8\times10^{-5}$ | | Suzuki et al. (1992) | Q | 232 |
| | $1.8\times10^{-5}$ | | Nirmalakhandan and Speece (1988) | Q | |
| | $1.6\times10^{-5}$ | | Yaws (1999) | ? | 21 |
| | $1.6\times10^{-5}$ | | Yaws and Yang (1992) | ? | 21 |
| 2-methyl-2-pentene $C_6H_{12}$ [625-27-4] JMMZCWZIJXAGKW-UHFFFAOYSA-N | $2.6\times10^{-5}$ | | Yaws (2003) | X | 237 |
| | $1.5\times10^{-5}$ | | Gharagheizi et al. (2012) | Q | |
| | $2.5\times10^{-5}$ | | Gharagheizi et al. (2010) | Q | 246 |
| | $7.9\times10^{-6}$ | | Modarresi et al. (2005) | Q | 247 |
| | $4.8\times10^{-5}$ | | Yao et al. (2002) | Q | 229, 267 |
| | $2.7\times10^{-5}$ | | Yaws (1999) | ? | 21 |
| *cis*-3-methyl-2-pentene $C_6H_{12}$ [922-62-3] BEQGRRJLJLVQAQ-XQRVVYSFSA-N | $2.6\times10^{-5}$ | | Yaws (2003) | X | 237 |
| | $1.6\times10^{-5}$ | | Gharagheizi et al. (2012) | Q | |
| | $2.5\times10^{-5}$ | | Gharagheizi et al. (2010) | Q | 246 |
| | $2.7\times10^{-5}$ | | Yaws (1999) | ? | 21 |
| *trans*-3-methyl-2-pentene $C_6H_{12}$ [616-12-6] BEQGRRJLJLVQAQ-GQCTYLIASA-N | $2.6\times10^{-5}$ | | Yaws (2003) | X | 237 |
| | $1.8\times10^{-5}$ | | Gharagheizi et al. (2012) | Q | |
| | $2.5\times10^{-5}$ | | Gharagheizi et al. (2010) | Q | 246 |
| | $5.5\times10^{-6}$ | | Modarresi et al. (2005) | Q | 247 |
| | $3.9\times10^{-5}$ | | Yao et al. (2002) | Q | 229 |
| | $2.6\times10^{-5}$ | | Yaws (1999) | ? | 21 |
| *cis*-4-methyl-2-pentene $C_6H_{12}$ [691-38-3] LGAQJENWWYGFSN-PLNGDYQASA-N | $2.8\times10^{-5}$ | | Yaws (2003) | X | 237 |
| | $1.7\times10^{-5}$ | | Gharagheizi et al. (2012) | Q | |
| | $2.8\times10^{-5}$ | | Gharagheizi et al. (2010) | Q | 246 |
| | $3.7\times10^{-6}$ | | Modarresi et al. (2005) | Q | 247 |
| | $3.3\times10^{-5}$ | | Yao et al. (2002) | Q | 229 |
| | $2.8\times10^{-5}$ | | Yaws (1999) | ? | 21 |
| *trans*-4-methyl-2-pentene $C_6H_{12}$ [674-76-0] LGAQJENWWYGFSN-SNAWJCMRSA-N | $2.8\times10^{-5}$ | | Yaws (2003) | X | 237 |
| | $1.8\times10^{-5}$ | | Gharagheizi et al. (2012) | Q | |
| | $2.8\times10^{-5}$ | | Gharagheizi et al. (2010) | Q | 246 |
| | $5.0\times10^{-6}$ | | Modarresi et al. (2005) | Q | 247 |
| | $3.1\times10^{-5}$ | | Yao et al. (2002) | Q | 229 |
| | $2.8\times10^{-5}$ | | Yaws (1999) | ? | 21 |



Table A2.3: Aliphatic alkenes and cycloalkenes (...continued)

| Substance<br>Formula<br>(Trivial Name)<br>[CAS Registry Number]<br><small>InChIKey</small> | $H_s^{cp}$<br>(at $T^\ominus$)<br>$\left[\dfrac{\mathrm{mol}}{\mathrm{m}^3\,\mathrm{Pa}}\right]$ | $\dfrac{\mathrm{d}\ln H_s^{cp}}{\mathrm{d}(1/T)}$<br><br>[K] | Reference | Type | Note |
|---|---|---|---|---|---|
| 3-methylenepentane<br>$C_6H_{12}$<br>(2-ethyl-1-butene)<br>[760-21-4]<br><small>RYKZRKKEYSRDNF-UHFFFAOYSA-N</small> | $2.7\times10^{-5}$<br>$2.0\times10^{-5}$<br>$2.3\times10^{-5}$<br>$3.9\times10^{-6}$<br>$4.3\times10^{-5}$<br>$2.7\times10^{-5}$ | | Yaws (2003)<br>Gharagheizi et al. (2012)<br>Gharagheizi et al. (2010)<br>Modarresi et al. (2005)<br>Yao et al. (2002)<br>Yaws (1999) | X<br>Q<br>Q<br>Q<br>Q<br>? | 237<br><br>246<br>247<br>229<br>21 |
| 2,3-dimethyl-1-butene<br>$C_6H_{12}$<br>[563-78-0]<br><small>OWWIWYDDISJUMY-UHFFFAOYSA-N</small> | $2.8\times10^{-5}$<br>$1.6\times10^{-5}$<br>$2.9\times10^{-5}$<br>$1.7\times10^{-5}$<br>$6.0\times10^{-6}$<br>$1.9\times10^{-5}$<br>$2.8\times10^{-5}$ | | Yaws (2003)<br>Gharagheizi et al. (2012)<br>Gharagheizi et al. (2010)<br>Hilal et al. (2008)<br>Modarresi et al. (2005)<br>Yao et al. (2002)<br>Yaws (1999) | X<br>Q<br>Q<br>Q<br>Q<br>Q<br>? | 237<br><br>246<br><br>247<br>229<br>21 |
| 3,3-dimethyl-1-butene<br>$C_6H_{12}$<br>[558-37-2]<br><small>PKXHXOTZMFCXSH-UHFFFAOYSA-N</small> | $2.8\times10^{-5}$<br>$1.6\times10^{-5}$<br>$2.6\times10^{-5}$<br>$5.2\times10^{-6}$<br>$1.0\times10^{-5}$<br>$2.9\times10^{-5}$ | | Yaws (2003)<br>Gharagheizi et al. (2012)<br>Gharagheizi et al. (2010)<br>Modarresi et al. (2005)<br>Yao et al. (2002)<br>Yaws (1999) | X<br>Q<br>Q<br>Q<br>Q<br>? | 237<br><br>246<br>247<br>229<br>21 |
| 2,3-dimethyl-2-butene<br>$C_6H_{12}$<br>[563-79-1]<br><small>WGLLSSPDPJPLOR-UHFFFAOYSA-N</small> | $2.5\times10^{-5}$<br>$1.4\times10^{-4}$<br>$1.1\times10^{-4}$<br>$5.5\times10^{-5}$<br>$1.2\times10^{-5}$<br>$2.7\times10^{-5}$<br>$4.9\times10^{-6}$<br>$2.7\times10^{-5}$<br>$2.6\times10^{-5}$ | | Yaws (2003)<br>Wang et al. (2017)<br>Wang et al. (2017)<br>Wang et al. (2017)<br>Gharagheizi et al. (2012)<br>Gharagheizi et al. (2010)<br>Modarresi et al. (2005)<br>Yao et al. (2002)<br>Yaws (1999) | X<br>Q<br>Q<br>Q<br>Q<br>Q<br>Q<br>Q<br>? | 237<br>80, 238<br>80, 239<br>80, 240<br><br>246<br>247<br>229<br>21 |
| 1-heptene<br>$C_7H_{14}$<br>[592-76-7]<br><small>ZGEGCLOFRBLKSE-UHFFFAOYSA-N</small> | $2.0\times10^{-5}$<br>$2.5\times10^{-5}$<br>$2.3\times10^{-5}$<br>$2.3\times10^{-5}$<br>$2.5\times10^{-5}$<br>$2.5\times10^{-5}$<br>$2.5\times10^{-5}$<br>$1.6\times10^{-3}$<br>$2.6\times10^{-5}$<br>$1.6\times10^{-5}$<br>$1.3\times10^{-5}$<br>$3.3\times10^{-5}$<br>$3.2\times10^{-6}$<br>$2.2\times10^{-5}$<br>$2.1\times10^{-5}$<br>$5.8\times10^{-5}$<br>$1.7\times10^{-5}$ | | Brockbank (2013)<br>Plyasunov and Shock (2000)<br>Duchowicz et al. (2020)<br>HSDB (2015)<br>Mackay et al. (2006a)<br>Mackay et al. (1993)<br>Yaws (2003)<br>Duchowicz et al. (2020)<br>Gharagheizi et al. (2012)<br>Gharagheizi et al. (2010)<br>Hilal et al. (2008)<br>Modarresi et al. (2007)<br>Modarresi et al. (2005)<br>Yao et al. (2002)<br>English and Carroll (2001)<br>Katritzky et al. (1998)<br>Nirmalakhandan et al. (1997) | L<br>L<br>V<br>V<br>V<br>V<br>X<br>Q<br>Q<br>Q<br>Q<br>Q<br>Q<br>Q<br>Q<br>Q<br>Q | <br><br>186<br><br><br><br>237<br><br><br>246<br><br>67<br>247<br>229<br>230, 231<br><br> |





Table A2.3: Aliphatic alkenes and cycloalkenes (. . . continued)

| Substance Formula (Trivial Name) [CAS Registry Number] InChIKey | $H_s^{cp}$ (at $T^{\ominus}$) $\left[\dfrac{\mathrm{mol}}{\mathrm{m^3\,Pa}}\right]$ | $\dfrac{\mathrm{d}\ln H_s^{cp}}{\mathrm{d}(1/T)}$ [K] | Reference | Type | Note |
|---|---|---|---|---|---|
| | $2.5\times10^{-5}$ | | Yaws (1999) | ? | 21 |
| | $2.5\times10^{-5}$ | | Yaws and Yang (1992) | ? | 21 |
| | $2.4\times10^{-5}$ | | Abraham et al. (1990) | ? | |
| 2-heptene C$_7$H$_{14}$ [592-77-8] OTTZHAVKAVGASB-UHFFFAOYSA-N | $2.3\times10^{-5}$ | | Plyasunov and Shock (2000) | L | |
| | $2.4\times10^{-5}$ | | Duchowicz et al. (2020) | V | 186 |
| | $5.2\times10^{-4}$ | | Duchowicz et al. (2020) | Q | |
| | $2.0\times10^{-5}$ | | Raventos-Duran et al. (2010) | Q | 271, 243 |
| | $2.0\times10^{-5}$ | | Raventos-Duran et al. (2010) | Q | 244 |
| | $1.6\times10^{-5}$ | | Raventos-Duran et al. (2010) | Q | 245 |
| | $1.7\times10^{-5}$ | | Hilal et al. (2008) | Q | |
| | $2.6\times10^{-5}$ | | Modarresi et al. (2007) | Q | 67 |
| | $2.1\times10^{-5}$ | | English and Carroll (2001) | Q | 230, 274 |
| *cis*-2-heptene C$_7$H$_{14}$ [6443-92-1] OTTZHAVKAVGASB-HYXAFXHYSA-N | $2.4\times10^{-5}$ | | Brockbank (2013) | L | |
| | $1.5\times10^{-5}$ | | Yaws (2003) | X | 237 |
| | $2.0\times10^{-5}$ | | Gharagheizi et al. (2012) | Q | |
| | $1.5\times10^{-5}$ | | Gharagheizi et al. (2010) | Q | 246 |
| | $9.1\times10^{-6}$ | | Modarresi et al. (2005) | Q | 247 |
| | $3.0\times10^{-5}$ | | Yao et al. (2002) | Q | 229, 267 |
| | $1.5\times10^{-5}$ | | Yaws (1999) | ? | 21 |
| *trans*-2-heptene C$_7$H$_{14}$ [14686-13-6] OTTZHAVKAVGASB-HWKANZROSA-N | $2.4\times10^{-5}$ | | Brockbank (2013) | L | |
| | $2.4\times10^{-5}$ | | Mackay and Shiu (1981) | L | |
| | $2.3\times10^{-5}$ | | Duchowicz et al. (2020) | V | 186 |
| | $2.4\times10^{-5}$ | | Mackay et al. (1993) | V | |
| | $2.4\times10^{-5}$ | | Eastcott et al. (1988) | V | |
| | $2.4\times10^{-5}$ | | Hine and Mookerjee (1975) | V | |
| | $1.5\times10^{-5}$ | | Yaws (2003) | X | 237 |
| | $5.2\times10^{-4}$ | | Duchowicz et al. (2020) | Q | |
| | $2.0\times10^{-5}$ | | Gharagheizi et al. (2012) | Q | |
| | $1.5\times10^{-5}$ | | Gharagheizi et al. (2010) | Q | 246 |
| | $2.5\times10^{-5}$ | | Modarresi et al. (2007) | Q | 67 |
| | $4.4\times10^{-6}$ | | Modarresi et al. (2005) | Q | 247 |
| | $2.8\times10^{-5}$ | | Yao et al. (2002) | Q | 229 |
| | $1.7\times10^{-5}$ | | Nirmalakhandan et al. (1997) | Q | |
| | $2.4\times10^{-5}$ | | Suzuki et al. (1992) | Q | 232 |
| | $1.5\times10^{-5}$ | | Yaws (1999) | ? | 21 |
| *cis*-3-heptene C$_7$H$_{14}$ [7642-10-6] WZHKDGJSXCTSCK-ALCCZGGFSA-N | $1.6\times10^{-5}$ | | Yaws (2003) | X | 237 |
| | $1.9\times10^{-5}$ | | Gharagheizi et al. (2012) | Q | |
| | $1.5\times10^{-5}$ | | Gharagheizi et al. (2010) | Q | 246 |
| | $3.8\times10^{-6}$ | | Modarresi et al. (2005) | Q | 247 |
| | $3.1\times10^{-5}$ | | Yao et al. (2002) | Q | 229 |
| | $1.6\times10^{-5}$ | | Yaws (1999) | ? | 21 |



Table A2.3: Aliphatic alkenes and cycloalkenes (…continued)

| Substance / Formula / (Trivial Name) / [CAS Registry Number] / InChIKey | $H_s^{cp}$ (at $T^{\ominus}$) $\left[\dfrac{\text{mol}}{\text{m}^3\,\text{Pa}}\right]$ | $\dfrac{\text{d}\ln H_s^{cp}}{\text{d}(1/T)}$ [K] | Reference | Type | Note |
|---|---|---|---|---|---|
| *trans*-3-heptene | $1.6\times10^{-5}$ | | Yaws (2003) | X | 237 |
| C$_7$H$_{14}$ | $1.9\times10^{-5}$ | | Gharagheizi et al. (2012) | Q | |
| [14686-14-7] | $1.5\times10^{-5}$ | | Gharagheizi et al. (2010) | Q | 246 |
| WZHKDGJSXCTSCK-FNORWQNLSA-N | $4.0\times10^{-6}$ | | Modarresi et al. (2005) | Q | 247 |
| | $2.8\times10^{-5}$ | | Yao et al. (2002) | Q | 229 |
| | $1.6\times10^{-5}$ | | Yaws (1999) | ? | 21 |
| 2-methyl-1-hexene | $1.7\times10^{-5}$ | | Yaws (2003) | X | 237 |
| C$_7$H$_{14}$ | $1.7\times10^{-5}$ | | Gharagheizi et al. (2012) | Q | |
| [6094-02-6] | $1.5\times10^{-5}$ | | Gharagheizi et al. (2010) | Q | 246 |
| IRUDSQHLKGNCGF-UHFFFAOYSA-N | $3.2\times10^{-6}$ | | Modarresi et al. (2005) | Q | 247 |
| | $2.4\times10^{-5}$ | | Yao et al. (2002) | Q | 229 |
| | $1.7\times10^{-5}$ | | Yaws (1999) | ? | 21 |
| 2-methyl-2-hexene | $1.6\times10^{-5}$ | | Yaws (2003) | X | 237 |
| C$_7$H$_{14}$ | $1.4\times10^{-5}$ | | Gharagheizi et al. (2012) | Q | |
| [2738-19-4] | $1.5\times10^{-5}$ | | Gharagheizi et al. (2010) | Q | 246 |
| BWEKDYGHDCHWEN-UHFFFAOYSA-N | | | | | |
| 2-methyl-*cis*-3-hexene | $1.9\times10^{-5}$ | | Yaws (2003) | X | 237 |
| C$_7$H$_{14}$ | $1.6\times10^{-5}$ | | Gharagheizi et al. (2012) | Q | |
| [15840-60-5] | $1.7\times10^{-5}$ | | Gharagheizi et al. (2010) | Q | 246 |
| IQANHWBWTVLDTP-WAYWQWQTSA-N | | | | | |
| 2-methyl-*trans*-3-hexene | $1.9\times10^{-5}$ | | Yaws (2003) | X | 237 |
| C$_7$H$_{14}$ | $1.6\times10^{-5}$ | | Gharagheizi et al. (2012) | Q | |
| [692-24-0] | $1.7\times10^{-5}$ | | Gharagheizi et al. (2010) | Q | 246 |
| IQANHWBWTVLDTP-AATRIKPKSA-N | | | | | |
| 3-methyl-1-hexene | $1.9\times10^{-5}$ | | Yaws (2003) | X | 237 |
| C$_7$H$_{14}$ | $2.1\times10^{-5}$ | | Gharagheizi et al. (2012) | Q | |
| [3404-61-3] | $1.7\times10^{-5}$ | | Gharagheizi et al. (2010) | Q | 246 |
| RITONZMLZWYPHW-UHFFFAOYSA-N | $9.9\times10^{-6}$ | | Modarresi et al. (2005) | Q | 247 |
| | $1.0\times10^{-5}$ | | Yao et al. (2002) | Q | 229 |
| | $1.9\times10^{-5}$ | | Yaws (1999) | ? | 21 |
| 3-methyl-*cis*-2-hexene | $1.6\times10^{-5}$ | | Yaws (2003) | X | 237 |
| C$_7$H$_{14}$ | $1.5\times10^{-5}$ | | Gharagheizi et al. (2012) | Q | |
| [10574-36-4] | $1.5\times10^{-5}$ | | Gharagheizi et al. (2010) | Q | 246 |
| JZMUUSXQSKCZNO-ALCCZGGFSA-N | | | | | |
| 3-methyl-*trans*-2-hexene | $1.6\times10^{-5}$ | | Yaws (2003) | X | 237 |
| C$_7$H$_{14}$ | $1.4\times10^{-5}$ | | Gharagheizi et al. (2012) | Q | |
| [20710-38-7] | $1.5\times10^{-5}$ | | Gharagheizi et al. (2010) | Q | 246 |
| JZMUUSXQSKCZNO-FNORWQNLSA-N | | | | | |
| 3-methyl-*cis*-3-hexene | $1.7\times10^{-5}$ | | Yaws (2003) | X | 237 |
| C$_7$H$_{14}$ | $1.4\times10^{-5}$ | | Gharagheizi et al. (2012) | Q | |
| [4914-89-0] | $1.5\times10^{-5}$ | | Gharagheizi et al. (2010) | Q | 246 |
| FHHSSXNRVNXTBG-SREVYHEPSA-N | | | | | |



Table A2.3: Aliphatic alkenes and cycloalkenes (... continued)

| Substance Formula (Trivial Name) [CAS Registry Number] InChIKey | $H_s^{cp}$ (at $T^\ominus$) $\left[\dfrac{\mathrm{mol}}{\mathrm{m^3\,Pa}}\right]$ | $\dfrac{\mathrm{d}\ln H_s^{cp}}{\mathrm{d}(1/T)}$ [K] | Reference | Type | Note |
|---|---|---|---|---|---|
| 3-methyl-*trans*-3-hexene | $1.7\times10^{-5}$ | | Yaws (2003) | X | 237 |
| C$_7$H$_{14}$ | $1.3\times10^{-5}$ | | Gharagheizi et al. (2012) | Q | |
| [3899-36-3] | $1.5\times10^{-5}$ | | Gharagheizi et al. (2010) | Q | 246 |
| FHHSSXNRVNXTBG-VOTSOKGWSA-N | | | | | |
| 4-methyl-1-hexene | $1.8\times10^{-5}$ | | Yaws (2003) | X | 237 |
| C$_7$H$_{14}$ | $2.4\times10^{-5}$ | | Gharagheizi et al. (2012) | Q | |
| [3769-23-1] | $1.7\times10^{-5}$ | | Gharagheizi et al. (2010) | Q | 246 |
| SUWJESCICIOQHO-UHFFFAOYSA-N | $4.8\times10^{-6}$ | | Modarresi et al. (2005) | Q | 247 |
| | $1.3\times10^{-5}$ | | Yao et al. (2002) | Q | 229, 267 |
| | $1.9\times10^{-5}$ | | Yaws (1999) | ? | 21 |
| 4-methyl-*cis*-2-hexene | $1.9\times10^{-5}$ | | Yaws (2003) | X | 237 |
| C$_7$H$_{14}$ | $1.6\times10^{-5}$ | | Gharagheizi et al. (2012) | Q | |
| [3683-19-0] | $1.7\times10^{-5}$ | | Gharagheizi et al. (2010) | Q | 246 |
| MBNDKEPQUVZHCM-XQRVVYSFSA-N | | | | | |
| 4-methyl-*trans*-2-hexene | $1.9\times10^{-5}$ | | Yaws (2003) | X | 237 |
| C$_7$H$_{14}$ | $1.6\times10^{-5}$ | | Gharagheizi et al. (2012) | Q | |
| [3683-22-5] | $1.7\times10^{-5}$ | | Gharagheizi et al. (2010) | Q | 246 |
| MBNDKEPQUVZHCM-GQCTYLIASA-N | | | | | |
| 5-methyl-1-hexene | $1.9\times10^{-5}$ | | Yaws (2003) | X | 237 |
| C$_7$H$_{14}$ | $2.2\times10^{-5}$ | | Gharagheizi et al. (2012) | Q | |
| [3524-73-0] | $1.7\times10^{-5}$ | | Gharagheizi et al. (2010) | Q | 246 |
| JIUFYGIESXPUPL-UHFFFAOYSA-N | | | | | |
| 5-methyl-*cis*-2-hexene | $1.8\times10^{-5}$ | | Yaws (2003) | X | 237 |
| C$_7$H$_{14}$ | $1.7\times10^{-5}$ | | Gharagheizi et al. (2012) | Q | |
| [13151-17-2] | $1.7\times10^{-5}$ | | Gharagheizi et al. (2010) | Q | 246 |
| GHBKCPRDHLITSE-PLNGDYQASA-N | | | | | |
| 5-methyl-*trans*-2-hexene | $1.8\times10^{-5}$ | | Yaws (2003) | X | 237 |
| C$_7$H$_{14}$ | $1.7\times10^{-5}$ | | Gharagheizi et al. (2012) | Q | |
| [7385-82-2] | $1.7\times10^{-5}$ | | Gharagheizi et al. (2010) | Q | 246 |
| GHBKCPRDHLITSE-SNAWJCMRSA-N | | | | | |
| 2,3-dimethyl-1-pentene | $1.9\times10^{-5}$ | | Yaws (2003) | X | 237 |
| C$_7$H$_{14}$ | $1.5\times10^{-5}$ | | Gharagheizi et al. (2012) | Q | |
| [3404-72-6] | $1.8\times10^{-5}$ | | Gharagheizi et al. (2010) | Q | 246 |
| LIMAEKMEXJTSNI-UHFFFAOYSA-N | | | | | |
| 2,3-dimethyl-2-pentene | $1.5\times10^{-5}$ | | Yaws (2003) | X | 237 |
| C$_7$H$_{14}$ | $1.0\times10^{-5}$ | | Gharagheizi et al. (2012) | Q | |
| [10574-37-5] | $1.5\times10^{-5}$ | | Gharagheizi et al. (2010) | Q | 246 |
| WFHALSLYRWWUGH-UHFFFAOYSA-N | | | | | |
| 2,4-dimethyl-1-pentene | $1.9\times10^{-5}$ | | Yaws (2003) | X | 237 |
| C$_7$H$_{14}$ | $1.4\times10^{-5}$ | | Gharagheizi et al. (2012) | Q | |
| [2213-32-3] | $1.8\times10^{-5}$ | | Gharagheizi et al. (2010) | Q | 246 |
| LXQPBCHJNIOMQU-UHFFFAOYSA-N | | | | | |



Table A2.3: Aliphatic alkenes and cycloalkenes (...continued)

| Substance Formula (Trivial Name) [CAS Registry Number] InChIKey | $H_s^{cp}$ (at $T^{\ominus}$) $\left[\dfrac{\mathrm{mol}}{\mathrm{m^3\,Pa}}\right]$ | $\dfrac{\mathrm{d}\ln H_s^{cp}}{\mathrm{d}(1/T)}$ [K] | Reference | Type | Note |
|---|---|---|---|---|---|
| 2,4-dimethyl-2-pentene C$_7$H$_{14}$ [625-65-0] VVCFYASOGFVJFN-UHFFFAOYSA-N | $2.0\times10^{-5}$ $9.9\times10^{-6}$ $1.9\times10^{-5}$ | | Yaws (2003) Gharagheizi et al. (2012) Gharagheizi et al. (2010) | X Q Q | 237  246 |
| 3,3-dimethyl-1-pentene C$_7$H$_{14}$ [3404-73-7] TXBZITDWMURSEF-UHFFFAOYSA-N | $2.1\times10^{-5}$ $1.8\times10^{-5}$ $1.8\times10^{-5}$ | | Yaws (2003) Gharagheizi et al. (2012) Gharagheizi et al. (2010) | X Q Q | 237  246 |
| 3,4-dimethyl-1-pentene C$_7$H$_{14}$ [7385-78-6] WFHXQNMTMDKVJG-UHFFFAOYSA-N | $2.0\times10^{-5}$ $2.0\times10^{-5}$ $1.9\times10^{-5}$ | | Yaws (2003) Gharagheizi et al. (2012) Gharagheizi et al. (2010) | X Q Q | 237  246 |
| 3,4-dimethyl-*cis*-2-pentene C$_7$H$_{14}$ [4914-91-4] PPBWEVVDSRKEIK-ALCCZGGFSA-N | $1.8\times10^{-5}$ $1.2\times10^{-5}$ $1.9\times10^{-5}$ | | Yaws (2003) Gharagheizi et al. (2012) Gharagheizi et al. (2010) | X Q Q | 237  246 |
| 3,4-dimethyl-*trans*-2-pentene C$_7$H$_{14}$ [4914-92-5] PPBWEVVDSRKEIK-FNORWQNLSA-N | $1.7\times10^{-5}$ $1.3\times10^{-5}$ $1.9\times10^{-5}$ | | Yaws (2003) Gharagheizi et al. (2012) Gharagheizi et al. (2010) | X Q Q | 237  246 |
| 4,4-dimethyl-1-pentene C$_7$H$_{14}$ [762-62-9] KLCNJIQZXOQYTE-UHFFFAOYSA-N | $2.1\times10^{-5}$ $1.6\times10^{-5}$ $1.8\times10^{-5}$ | | Yaws (2003) Gharagheizi et al. (2012) Gharagheizi et al. (2010) | X Q Q | 237  246 |
| 4,4-dimethyl-*cis*-2-pentene C$_7$H$_{14}$ [762-63-0] BIDIHFPLDRSAMB-WAYWQWQTSA-N | $1.9\times10^{-5}$ $1.4\times10^{-5}$ $1.8\times10^{-5}$ | | Yaws (2003) Gharagheizi et al. (2012) Gharagheizi et al. (2010) | X Q Q | 237  246 |
| 4,4-dimethyl-*trans*-2-pentene C$_7$H$_{14}$ [690-08-4] BIDIHFPLDRSAMB-AATRIKPKSA-N | $2.1\times10^{-5}$ $1.2\times10^{-5}$ $1.8\times10^{-5}$ | | Yaws (2003) Gharagheizi et al. (2012) Gharagheizi et al. (2010) | X Q Q | 237  246 |
| 2-ethyl-1-pentene C$_7$H$_{14}$ [3404-71-5] TWCRBBJSQAZZQB-UHFFFAOYSA-N | $1.6\times10^{-5}$ $1.9\times10^{-5}$ $1.5\times10^{-5}$ $2.9\times10^{-6}$ $2.4\times10^{-5}$ $1.6\times10^{-5}$ | | Yaws (2003) Gharagheizi et al. (2012) Gharagheizi et al. (2010) Modarresi et al. (2005) Yao et al. (2002) Yaws (1999) | X Q Q Q Q ? | 237  246 247 229 21 |



Table A2.3: Aliphatic alkenes and cycloalkenes (...continued)

| Substance<br>Formula<br>(Trivial Name)<br>[CAS Registry Number]<br>InChIKey | $H_s^{cp}$<br>(at $T^\ominus$)<br>$\left[\dfrac{\text{mol}}{\text{m}^3\,\text{Pa}}\right]$ | $\dfrac{\text{d}\ln H_s^{cp}}{\text{d}(1/T)}$<br><br>[K] | Reference | Type | Note |
|---|---|---|---|---|---|
| 3-ethyl-1-pentene<br>C$_7$H$_{14}$<br>[4038-04-4]<br>YPVPQMCSLFDIKA-UHFFFAOYSA-N | $1.9\times10^{-5}$<br>$2.2\times10^{-5}$<br>$1.7\times10^{-5}$<br>$4.9\times10^{-6}$<br>$1.3\times10^{-5}$<br>$1.9\times10^{-5}$ | | Yaws (2003)<br>Gharagheizi et al. (2012)<br>Gharagheizi et al. (2010)<br>Modarresi et al. (2005)<br>Yao et al. (2002)<br>Yaws (1999) | X<br>Q<br>Q<br>Q<br>Q<br>? | 237<br><br>246<br>247<br>229<br>21 |
| 3-ethyl-2-pentene<br>C$_7$H$_{14}$<br>[816-79-5]<br>XMYFZAWUNVHVGI-UHFFFAOYSA-N | $1.6\times10^{-5}$<br>$1.4\times10^{-5}$<br>$1.5\times10^{-5}$ | | Yaws (2003)<br>Gharagheizi et al. (2012)<br>Gharagheizi et al. (2010) | X<br>Q<br>Q | 237<br><br>246 |
| 3-methyl-2-ethyl-1-butene<br>C$_7$H$_{14}$<br>[7357-93-9]<br>ADHCYQWFCLQBFG-UHFFFAOYSA-N | $1.9\times10^{-5}$<br>$1.6\times10^{-5}$<br>$1.8\times10^{-5}$ | | Yaws (2003)<br>Gharagheizi et al. (2012)<br>Gharagheizi et al. (2010) | X<br>Q<br>Q | 237<br><br>246 |
| 2,3,3-trimethyl-1-butene<br>C$_7$H$_{14}$<br>[594-56-9]<br>AUYRUAVCWOAHQN-UHFFFAOYSA-N | $1.9\times10^{-5}$<br>$1.4\times10^{-5}$<br>$1.9\times10^{-5}$<br>$3.0\times10^{-6}$<br>$7.8\times10^{-6}$<br>$2.0\times10^{-5}$ | | Yaws (2003)<br>Gharagheizi et al. (2012)<br>Gharagheizi et al. (2010)<br>Modarresi et al. (2005)<br>Yao et al. (2002)<br>Yaws (1999) | X<br>Q<br>Q<br>Q<br>Q<br>? | 237<br><br>246<br>247<br>229<br>21 |
| 1-octene<br>C$_8$H$_{16}$<br>[111-66-0]<br>KWKAKUADMBZCLK-UHFFFAOYSA-N | $1.6\times10^{-5}$<br>$1.1\times10^{-5}$<br>$1.0\times10^{-5}$<br>$1.6\times10^{-5}$<br>$1.6\times10^{-5}$<br>$1.0\times10^{-5}$<br>$1.0\times10^{-5}$<br>$1.0\times10^{-5}$<br>$1.0\times10^{-5}$<br>$1.0\times10^{-5}$<br>$1.0\times10^{-5}$<br>$1.5\times10^{-5}$<br>$1.6\times10^{-5}$<br>$1.6\times10^{-3}$<br>$2.2\times10^{-5}$<br>$1.6\times10^{-5}$<br>$9.9\times10^{-6}$<br>$1.6\times10^{-5}$<br>$1.2\times10^{-5}$<br>$9.2\times10^{-6}$<br>$2.6\times10^{-5}$<br>$2.1\times10^{-6}$<br>$1.1\times10^{-5}$<br>$1.9\times10^{-5}$<br>$1.6\times10^{-5}$ | 4400 | Brockbank (2013)<br>Plyasunov and Shock (2000)<br>Mackay and Shiu (1981)<br>Duchowicz et al. (2020)<br>HSDB (2015)<br>Mackay et al. (2006a)<br>Mackay et al. (1993)<br>Hwang et al. (1992)<br>Meylan and Howard (1991)<br>Eastcott et al. (1988)<br>Hine and Mookerjee (1975)<br>McAuliffe (1966)<br>Yaws (2003)<br>Duchowicz et al. (2020)<br>Gharagheizi et al. (2012)<br>Raventos-Duran et al. (2010)<br>Raventos-Duran et al. (2010)<br>Raventos-Duran et al. (2010)<br>Gharagheizi et al. (2010)<br>Hilal et al. (2008)<br>Modarresi et al. (2007)<br>Modarresi et al. (2005)<br>Yaffe et al. (2003)<br>Yao et al. (2002)<br>English and Carroll (2001) | L<br>L<br>L<br>V<br>V<br>V<br>V<br>V<br>V<br>V<br>V<br>V<br>X<br>Q<br>Q<br>Q<br>Q<br>Q<br>Q<br>Q<br>Q<br>Q<br>Q<br>Q<br>Q | <br><br><br>186<br><br><br><br><br><br><br><br>24<br>237<br><br><br>242, 243<br>244<br>245<br>246<br><br>67<br>247<br>248, 249<br>229<br>230, 231 |



Table A2.3: Aliphatic alkenes and cycloalkenes (...continued)

| Substance Formula (Trivial Name) [CAS Registry Number] InChIKey | $H_s^{cp}$ (at $T^{\ominus}$) $\left[\dfrac{\mathrm{mol}}{\mathrm{m}^3\,\mathrm{Pa}}\right]$ | $\dfrac{\mathrm{d}\ln H_s^{cp}}{\mathrm{d}(1/T)}$ [K] | Reference | Type | Note |
|---|---|---|---|---|---|
| | $5.7\times10^{-5}$ | | Katritzky et al. (1998) | Q | |
| | $8.4\times10^{-6}$ | | Russell et al. (1992) | Q | 279 |
| | $1.8\times10^{-5}$ | | Suzuki et al. (1992) | Q | 232 |
| | $1.6\times10^{-5}$ | | Meylan and Howard (1991) | Q | |
| | $1.3\times10^{-5}$ | | Nirmalakhandan and Speece (1988) | Q | |
| | $1.6\times10^{-5}$ | | Yaws (1999) | ? | 21 |
| | $1.6\times10^{-5}$ | | Yaws and Yang (1992) | ? | 21 |
| | $1.6\times10^{-5}$ | | Abraham et al. (1990) | ? | |
| *cis*-2-octene $C_8H_{16}$ [7642-04-8] ILPBINAXDRFYPL-HYXAFXHYSA-N | $8.3\times10^{-6}$ $1.9\times10^{-5}$ $9.6\times10^{-6}$ | | Yaws (2003) Gharagheizi et al. (2012) Gharagheizi et al. (2010) | X Q Q | 237 246 |
| *trans*-2-octene $C_8H_{16}$ [13389-42-9] ILPBINAXDRFYPL-HWKANZROSA-N | $8.3\times10^{-6}$ $1.7\times10^{-5}$ $9.6\times10^{-6}$ $8.4\times10^{-6}$ | | Yaws (2003) Gharagheizi et al. (2012) Gharagheizi et al. (2010) Yaws (1999) | X Q Q ? | 237 246 21 |
| *cis*-3-octene $C_8H_{16}$ [14850-22-7] YCTDZYMMFQCTEO-ALCCZGGFSA-N | $8.9\times10^{-6}$ $1.7\times10^{-5}$ $9.6\times10^{-6}$ | | Yaws (2003) Gharagheizi et al. (2012) Gharagheizi et al. (2010) | X Q Q | 237 246 |
| *trans*-3-octene $C_8H_{16}$ [14919-01-8] YCTDZYMMFQCTEO-FNORWQNLSA-N | $8.6\times10^{-6}$ $1.6\times10^{-5}$ $9.6\times10^{-6}$ $8.8\times10^{-6}$ | | Yaws (2003) Gharagheizi et al. (2012) Gharagheizi et al. (2010) Yaws (1999) | X Q Q ? | 237 246 21 |
| *cis*-4-octene $C_8H_{16}$ [7642-15-1] IRUCBBFNLDIMIK-FPLPWBNLSA-N | $8.9\times10^{-6}$ $1.7\times10^{-5}$ $9.6\times10^{-6}$ | | Yaws (2003) Gharagheizi et al. (2012) Gharagheizi et al. (2010) | X Q Q | 237 246 |
| *trans*-4-octene $C_8H_{16}$ [14850-23-8] IRUCBBFNLDIMIK-BQYQJAHWSA-N | $8.9\times10^{-6}$ $1.6\times10^{-5}$ $9.6\times10^{-6}$ $9.1\times10^{-6}$ | | Yaws (2003) Gharagheizi et al. (2012) Gharagheizi et al. (2010) Yaws (1999) | X Q Q ? | 237 246 21 |
| 2-methyl-1-heptene $C_8H_{16}$ [15870-10-7] RCBGGJURENJHKV-UHFFFAOYSA-N | $9.6\times10^{-6}$ $1.5\times10^{-5}$ $9.8\times10^{-6}$ | | Yaws (2003) Gharagheizi et al. (2012) Gharagheizi et al. (2010) | X Q Q | 237 246 |
| 2-methyl-2-heptene $C_8H_{16}$ [627-97-4] WEPNJTDVIIKRIK-UHFFFAOYSA-N | $8.9\times10^{-6}$ $1.2\times10^{-5}$ $9.1\times10^{-6}$ | | Yaws (2003) Gharagheizi et al. (2012) Gharagheizi et al. (2010) | X Q Q | 237 246 |



Table A2.3: Aliphatic alkenes and cycloalkenes (...continued)

| Substance<br>Formula<br>(Trivial Name)<br>[CAS Registry Number]<br>InChIKey | $H_s^{cp}$<br>(at $T^{\ominus}$)<br>$\left[\dfrac{\mathrm{mol}}{\mathrm{m^3\,Pa}}\right]$ | $\dfrac{\mathrm{d\ln} H_s^{cp}}{\mathrm{d}(1/T)}$<br>[K] | Reference | Type | Note |
|---|---|---|---|---|---|
| 2-methyl-*cis*-3-heptene<br>C$_8$H$_{16}$<br>[20488-34-0]<br>CYEZJYAMLNTSKN-SREVYHEPSA-N | $1.1\times10^{-5}$<br>$1.3\times10^{-5}$<br>$1.1\times10^{-5}$ | | Yaws (2003)<br>Gharagheizi et al. (2012)<br>Gharagheizi et al. (2010) | X<br>Q<br>Q | 237<br><br>246 |
| 2-methyl-*trans*-3-heptene<br>C$_8$H$_{16}$<br>[692-96-6]<br>CYEZJYAMLNTSKN-VOTSOKGWSA-N | $1.1\times10^{-5}$<br>$1.3\times10^{-5}$<br>$1.1\times10^{-5}$ | | Yaws (2003)<br>Gharagheizi et al. (2012)<br>Gharagheizi et al. (2010) | X<br>Q<br>Q | 237<br><br>246 |
| 3-methyl-1-heptene<br>C$_8$H$_{16}$<br>[4810-09-7]<br>QDMFTFWKTYXBIW-UHFFFAOYSA-N | $1.1\times10^{-5}$<br>$1.8\times10^{-5}$<br>$1.2\times10^{-5}$ | | Yaws (2003)<br>Gharagheizi et al. (2012)<br>Gharagheizi et al. (2010) | X<br>Q<br>Q | 237<br><br>246 |
| 3-methyl-*cis*-2-heptene<br>C$_8$H$_{16}$<br>[22768-19-0]<br>OFKLSPUVNMOIJB-YVMONPNESA-N | $9.0\times10^{-6}$<br>$1.1\times10^{-5}$<br>$9.1\times10^{-6}$ | | Yaws (2003)<br>Gharagheizi et al. (2012)<br>Gharagheizi et al. (2010) | X<br>Q<br>Q | 237<br><br>246 |
| 3-methyl-*trans*-2-heptene<br>C$_8$H$_{16}$<br>[22768-20-3]<br>OFKLSPUVNMOIJB-VMPITWQZSA-N | $9.0\times10^{-6}$<br>$1.1\times10^{-5}$<br>$9.1\times10^{-6}$ | | Yaws (2003)<br>Gharagheizi et al. (2012)<br>Gharagheizi et al. (2010) | X<br>Q<br>Q | 237<br><br>246 |
| 3-methyl-*cis*-3-heptene<br>C$_8$H$_{16}$<br>[22768-17-8]<br>AAUHUDBDDBJONC-FPLPWBNLSA-N | $9.2\times10^{-6}$<br>$1.1\times10^{-5}$<br>$9.1\times10^{-6}$ | | Yaws (2003)<br>Gharagheizi et al. (2012)<br>Gharagheizi et al. (2010) | X<br>Q<br>Q | 237<br><br>246 |
| 3-methyl-*trans*-3-heptene<br>C$_8$H$_{16}$<br>[22768-18-9]<br>AAUHUDBDDBJONC-BQYQJAHWSA-N | $9.2\times10^{-6}$<br>$1.1\times10^{-5}$<br>$9.1\times10^{-6}$ | | Yaws (2003)<br>Gharagheizi et al. (2012)<br>Gharagheizi et al. (2010) | X<br>Q<br>Q | 237<br><br>246 |
| 4-methyl-1-heptene<br>C$_8$H$_{16}$<br>[13151-05-8]<br>BFGOGLKYJXQPJZ-UHFFFAOYSA-N | $1.1\times10^{-5}$<br>$1.9\times10^{-5}$<br>$1.2\times10^{-5}$ | | Yaws (2003)<br>Gharagheizi et al. (2012)<br>Gharagheizi et al. (2010) | X<br>Q<br>Q | 237<br><br>246 |
| 4-methyl-*cis*-2-heptene<br>C$_8$H$_{16}$<br>[66225-16-9]<br>SVGLFIBXFVQUQY-XQRVVYSFSA-N | $1.1\times10^{-5}$<br>$1.3\times10^{-5}$<br>$1.1\times10^{-5}$ | | Yaws (2003)<br>Gharagheizi et al. (2012)<br>Gharagheizi et al. (2010) | X<br>Q<br>Q | 237<br><br>246 |
| 4-methyl-*trans*-2-heptene<br>C$_8$H$_{16}$<br>[66225-17-0]<br>SVGLFIBXFVQUQY-GQCTYLIASA-N | $1.1\times10^{-5}$<br>$1.3\times10^{-5}$<br>$1.1\times10^{-5}$ | | Yaws (2003)<br>Gharagheizi et al. (2012)<br>Gharagheizi et al. (2010) | X<br>Q<br>Q | 237<br><br>246 |



Table A2.3: Aliphatic alkenes and cycloalkenes (...continued)

| Substance<br>Formula<br>(Trivial Name)<br>[CAS Registry Number]<br>InChIKey | $H_s^{cp}$<br>(at $T^{\ominus}$)<br>$\left[\dfrac{\mathrm{mol}}{\mathrm{m^3\,Pa}}\right]$ | $\dfrac{\mathrm{d}\ln H_s^{cp}}{\mathrm{d}(1/T)}$<br><br>[K] | Reference | Type | Note |
|---|---|---|---|---|---|
| 4-methyl-*cis*-3-heptene<br>C$_8$H$_{16}$<br>[14255-24-4]<br>KKVVJQGDNYIIMN-VURMDHGXSA-N | $9.0\times10^{-6}$<br>$1.1\times10^{-5}$<br>$9.1\times10^{-6}$ | | Yaws (2003)<br>Gharagheizi et al. (2012)<br>Gharagheizi et al. (2010) | X<br>Q<br>Q | 237<br><br>246 |
| 4-methyl-*trans*-3-heptene<br>C$_8$H$_{16}$<br>[13714-85-7]<br>KKVVJQGDNYIIMN-SOFGYWHQSA-N | $9.0\times10^{-6}$<br>$1.1\times10^{-5}$<br>$9.1\times10^{-6}$ | | Yaws (2003)<br>Gharagheizi et al. (2012)<br>Gharagheizi et al. (2010) | X<br>Q<br>Q | 237<br><br>246 |
| 5-methyl-1-heptene<br>C$_8$H$_{16}$<br>[13151-04-7]<br>WNEYWVBECXCQRT-UHFFFAOYSA-N | $1.1\times10^{-5}$<br>$1.9\times10^{-5}$<br>$1.2\times10^{-5}$ | | Yaws (2003)<br>Gharagheizi et al. (2012)<br>Gharagheizi et al. (2010) | X<br>Q<br>Q | 237<br><br>246 |
| 5-methyl-*cis*-2-heptene<br>C$_8$H$_{16}$<br>[24608-84-2]<br>VIHUHUGDEZCPDK-XQRVVYSFSA-N | $9.9\times10^{-6}$<br>$1.5\times10^{-5}$<br>$1.1\times10^{-5}$ | | Yaws (2003)<br>Gharagheizi et al. (2012)<br>Gharagheizi et al. (2010) | X<br>Q<br>Q | 237<br><br>246 |
| 5-methyl-*trans*-2-heptene<br>C$_8$H$_{16}$<br>[24608-85-3]<br>VIHUHUGDEZCPDK-GQCTYLIASA-N | $9.9\times10^{-6}$<br>$1.5\times10^{-5}$<br>$1.1\times10^{-5}$ | | Yaws (2003)<br>Gharagheizi et al. (2012)<br>Gharagheizi et al. (2010) | X<br>Q<br>Q | 237<br><br>246 |
| 5-methyl-*cis*-3-heptene<br>C$_8$H$_{16}$<br>[50422-80-5]<br>YMNTZRCUPAYGLG-SREVYHEPSA-N | $1.1\times10^{-5}$<br>$1.3\times10^{-5}$<br>$1.1\times10^{-5}$ | | Yaws (2003)<br>Gharagheizi et al. (2012)<br>Gharagheizi et al. (2010) | X<br>Q<br>Q | 237<br><br>246 |
| 5-methyl-*trans*-3-heptene<br>C$_8$H$_{16}$<br>[53510-18-2]<br>YMNTZRCUPAYGLG-VOTSOKGWSA-N | $1.1\times10^{-5}$<br>$1.3\times10^{-5}$<br>$1.1\times10^{-5}$ | | Yaws (2003)<br>Gharagheizi et al. (2012)<br>Gharagheizi et al. (2010) | X<br>Q<br>Q | 237<br><br>246 |
| 6-methyl-1-heptene<br>C$_8$H$_{16}$<br>[5026-76-6]<br>DFVOXRAAHOJJBN-UHFFFAOYSA-N | $1.1\times10^{-5}$<br>$1.9\times10^{-5}$<br>$1.2\times10^{-5}$ | | Yaws (2003)<br>Gharagheizi et al. (2012)<br>Gharagheizi et al. (2010) | X<br>Q<br>Q | 237<br><br>246 |
| 6-methyl-*cis*-2-heptene<br>C$_8$H$_{16}$<br>[66225-18-1]<br>LXBJRNXXTAWCKU-PLNGDYQASA-N | $1.0\times10^{-5}$<br>$1.5\times10^{-5}$<br>$1.1\times10^{-5}$ | | Yaws (2003)<br>Gharagheizi et al. (2012)<br>Gharagheizi et al. (2010) | X<br>Q<br>Q | 237<br><br>246 |
| 6-methyl-*trans*-2-heptene<br>C$_8$H$_{16}$<br>[51065-65-7]<br>LXBJRNXXTAWCKU-SNAWJCMRSA-N | $1.0\times10^{-5}$<br>$1.5\times10^{-5}$<br>$1.1\times10^{-5}$ | | Yaws (2003)<br>Gharagheizi et al. (2012)<br>Gharagheizi et al. (2010) | X<br>Q<br>Q | 237<br><br>246 |





Table A2.3: Aliphatic alkenes and cycloalkenes (... continued)

| Substance Formula (Trivial Name) [CAS Registry Number] InChIKey | $H_s^{cp}$ (at $T^{\ominus}$) $\left[\dfrac{\mathrm{mol}}{\mathrm{m^3\,Pa}}\right]$ | $\dfrac{\mathrm{d}\ln H_s^{cp}}{\mathrm{d}(1/T)}$ [K] | Reference | Type | Note |
|---|---|---|---|---|---|
| 6-methyl-*cis*-3-heptene C$_8$H$_{16}$ [66225-19-2] PMPISKBGRHSPEE-WAYWQWQTSA-N | $1.1\times10^{-5}$ $1.4\times10^{-5}$ $1.1\times10^{-5}$ | | Yaws (2003) Gharagheizi et al. (2012) Gharagheizi et al. (2010) | X Q Q | 237 246 |
| 6-methyl-*trans*-3-heptene C$_8$H$_{16}$ [66225-20-5] PMPISKBGRHSPEE-AATRIKPKSA-N | $1.0\times10^{-5}$ $1.4\times10^{-5}$ $1.1\times10^{-5}$ | | Yaws (2003) Gharagheizi et al. (2012) Gharagheizi et al. (2010) | X Q Q | 237 246 |
| 2,2-dimethyl-*cis*-3-hexene C$_8$H$_{16}$ [690-92-6] JPLZSSHKQZJYTJ-SREVYHEPSA-N | $1.2\times10^{-5}$ $1.1\times10^{-5}$ $1.2\times10^{-5}$ | | Yaws (2003) Gharagheizi et al. (2012) Gharagheizi et al. (2010) | X Q Q | 237 246 |
| 2,2-dimethyl-*trans*-3-hexene C$_8$H$_{16}$ [690-93-7] JPLZSSHKQZJYTJ-VOTSOKGWSA-N | $1.4\times10^{-5}$ $9.4\times10^{-6}$ $1.2\times10^{-5}$ | | Yaws (2003) Gharagheizi et al. (2012) Gharagheizi et al. (2010) | X Q Q | 237 246 |
| 2,3-dimethyl-1-hexene C$_8$H$_{16}$ [16746-86-4] LVLXQRZPKUFJJQ-UHFFFAOYSA-N | $1.2\times10^{-5}$ $1.2\times10^{-5}$ $1.1\times10^{-5}$ | | Yaws (2003) Gharagheizi et al. (2012) Gharagheizi et al. (2010) | X Q Q | 237 246 |
| 2,3-dimethyl-2-hexene C$_8$H$_{16}$ [7145-20-2] RGYAVZGBAJFMIZ-UHFFFAOYSA-N | $9.1\times10^{-6}$ $7.8\times10^{-6}$ $9.0\times10^{-6}$ | | Yaws (2003) Gharagheizi et al. (2012) Gharagheizi et al. (2010) | X Q Q | 237 246 |
| 2,3-dimethyl-*cis*-3-hexene C$_8$H$_{16}$ [59643-75-3] PRTXQHCLTIKAAJ-VURMDHGXSA-N | $1.1\times10^{-5}$ $9.2\times10^{-6}$ $1.1\times10^{-5}$ | | Yaws (2003) Gharagheizi et al. (2012) Gharagheizi et al. (2010) | X Q Q | 237 246 |
| 2,3-dimethyl-*trans*-3-hexene C$_8$H$_{16}$ [66225-30-7] PRTXQHCLTIKAAJ-SOFGYWHQSA-N | $1.1\times10^{-5}$ $9.2\times10^{-6}$ $1.1\times10^{-5}$ | | Yaws (2003) Gharagheizi et al. (2012) Gharagheizi et al. (2010) | X Q Q | 237 246 |
| 2,4-dimethyl-1-hexene C$_8$H$_{16}$ [16746-87-5] PKVDGQHNRICJLA-UHFFFAOYSA-N | $1.1\times10^{-5}$ $1.2\times10^{-5}$ $1.1\times10^{-5}$ | | Yaws (2003) Gharagheizi et al. (2012) Gharagheizi et al. (2010) | X Q Q | 237 246 |
| 2,4-dimethyl-2-hexene C$_8$H$_{16}$ [14255-23-3] IZSBIPQYBGDXJZ-UHFFFAOYSA-N | $1.2\times10^{-5}$ $8.4\times10^{-6}$ $1.1\times10^{-5}$ | | Yaws (2003) Gharagheizi et al. (2012) Gharagheizi et al. (2010) | X Q Q | 237 246 |



Table A2.3: Aliphatic alkenes and cycloalkenes (...continued)

| Substance Formula (Trivial Name) [CAS Registry Number] InChIKey | $H_s^{cp}$ (at $T^{\ominus}$) $\left[\dfrac{\text{mol}}{\text{m}^3\,\text{Pa}}\right]$ | $\dfrac{\text{d}\ln H_s^{cp}}{\text{d}(1/T)}$ [K] | Reference | Type | Note |
|---|---|---|---|---|---|
| 2,4-dimethyl-*cis*-3-hexene C$_8$H$_{16}$ [37549-89-6] VFCHHMABGOYOQI-VURMDHGXSA-N | $1.2\times10^{-5}$ $8.0\times10^{-6}$ $1.1\times10^{-5}$ | | Yaws (2003) Gharagheizi et al. (2012) Gharagheizi et al. (2010) | X Q Q | 237 246 |
| 2,4-dimethyl-*trans*-3-hexene C$_8$H$_{16}$ [61847-78-7] VFCHHMABGOYOQI-SOFGYWHQSA-N | $1.2\times10^{-5}$ $7.7\times10^{-6}$ $1.1\times10^{-5}$ | | Yaws (2003) Gharagheizi et al. (2012) Gharagheizi et al. (2010) | X Q Q | 237 246 |
| 2,5-dimethyl-1-hexene C$_8$H$_{16}$ [6975-92-4] ISZWTVCVSJVEOL-UHFFFAOYSA-N | $1.1\times10^{-5}$ $1.2\times10^{-5}$ $1.1\times10^{-5}$ | | Yaws (2003) Gharagheizi et al. (2012) Gharagheizi et al. (2010) | X Q Q | 237 246 |
| 2,5-dimethyl-2-hexene C$_8$H$_{16}$ [3404-78-2] VFZIUYUUQFYZBR-UHFFFAOYSA-N | $1.2\times10^{-5}$ $8.8\times10^{-6}$ $1.1\times10^{-5}$ | | Yaws (2003) Gharagheizi et al. (2012) Gharagheizi et al. (2010) | X Q Q | 237 246 |
| 2,5-dimethyl-*cis*-3-hexene C$_8$H$_{16}$ [10557-44-5] KNCMKWVOMRUHKZ-WAYWQWQTSA-N | $1.3\times10^{-5}$ $9.6\times10^{-6}$ $1.3\times10^{-5}$ | | Yaws (2003) Gharagheizi et al. (2012) Gharagheizi et al. (2010) | X Q Q | 237 246 |
| 2,5-dimethyl-*trans*-3-hexene C$_8$H$_{16}$ [692-70-6] KNCMKWVOMRUHKZ-AATRIKPKSA-N | $1.4\times10^{-5}$ $9.6\times10^{-6}$ $1.3\times10^{-5}$ | | Yaws (2003) Gharagheizi et al. (2012) Gharagheizi et al. (2010) | X Q Q | 237 246 |
| 3,3-dimethyl-1-hexene C$_8$H$_{16}$ [3404-77-1] RXYYKIMRVXDSFR-UHFFFAOYSA-N | $1.3\times10^{-5}$ $1.5\times10^{-5}$ $1.3\times10^{-5}$ | | Yaws (2003) Gharagheizi et al. (2012) Gharagheizi et al. (2010) | X Q Q | 237 246 |
| 3,4-dimethyl-1-hexene C$_8$H$_{16}$ [16745-94-1] OWWRMMIWAOBBFK-UHFFFAOYSA-N | $1.1\times10^{-5}$ $1.9\times10^{-5}$ $1.3\times10^{-5}$ | | Yaws (2003) Gharagheizi et al. (2012) Gharagheizi et al. (2010) | X Q Q | 237 246 |
| 3,4-dimethyl-*cis*-2-hexene C$_8$H$_{16}$ [19550-81-3] FMNLVKMLDPGPRY-ALCCZGGFSA-N | $1.0\times10^{-5}$ $9.8\times10^{-6}$ $1.1\times10^{-5}$ | | Yaws (2003) Gharagheizi et al. (2012) Gharagheizi et al. (2010) | X Q Q | 237 246 |
| 3,4-dimethyl-*trans*-2-hexene C$_8$H$_{16}$ [19550-82-4] FMNLVKMLDPGPRY-FNORWQNLSA-N | $1.0\times10^{-5}$ $9.8\times10^{-6}$ $1.1\times10^{-5}$ | | Yaws (2003) Gharagheizi et al. (2012) Gharagheizi et al. (2010) | X Q Q | 237 246 |



Table A2.3: Aliphatic alkenes and cycloalkenes (…continued)

| Substance<br>Formula<br>(Trivial Name)<br>[CAS Registry Number]<br>InChIKey | $H_s^{cp}$ (at $T^{\ominus}$) $\left[\dfrac{\text{mol}}{\text{m}^3\,\text{Pa}}\right]$ | $\dfrac{\text{d}\ln H_s^{cp}}{\text{d}(1/T)}$ [K] | Reference | Type | Note |
|---|---|---|---|---|---|
| 3,4-dimethyl-*cis*-3-hexene<br>C$_8$H$_{16}$<br>[19550-87-9]<br>XTUXVDJHGIEBAA-FPLPWBNLSA-N | $9.0\times10^{-6}$<br>$7.9\times10^{-6}$<br>$9.0\times10^{-6}$ | | Yaws (2003)<br>Gharagheizi et al. (2012)<br>Gharagheizi et al. (2010) | X<br>Q<br>Q | 237<br><br>246 |
| 3,4-dimethyl-*trans*-3-hexene<br>C$_8$H$_{16}$<br>[19550-88-0]<br>XTUXVDJHGIEBAA-BQYQJAHWSA-N | $9.0\times10^{-6}$<br>$7.9\times10^{-6}$<br>$9.0\times10^{-6}$ | | Yaws (2003)<br>Gharagheizi et al. (2012)<br>Gharagheizi et al. (2010) | X<br>Q<br>Q | 237<br><br>246 |
| 3,5-dimethyl-1-hexene<br>C$_8$H$_{16}$<br>[7423-69-0]<br>FEZKAPRRVNNJTK-UHFFFAOYSA-N | $1.3\times10^{-5}$<br>$1.5\times10^{-5}$<br>$1.3\times10^{-5}$ | | Yaws (2003)<br>Gharagheizi et al. (2012)<br>Gharagheizi et al. (2010) | X<br>Q<br>Q | 237<br><br>246 |
| 3,5-dimethyl-*cis*-2-hexene<br>C$_8$H$_{16}$<br>[66225-31-8]<br>BQAZYKYBFAMHPG-YVMONPNESA-N | $1.1\times10^{-5}$<br>$8.7\times10^{-6}$<br>$1.1\times10^{-5}$ | | Yaws (2003)<br>Gharagheizi et al. (2012)<br>Gharagheizi et al. (2010) | X<br>Q<br>Q | 237<br><br>246 |
| 3,5-dimethyl-*trans*-2-hexene<br>C$_8$H$_{16}$<br>[66225-12-5]<br>BQAZYKYBFAMHPG-VMPITWQZSA-N | $1.1\times10^{-5}$<br>$8.7\times10^{-6}$<br>$1.1\times10^{-5}$ | | Yaws (2003)<br>Gharagheizi et al. (2012)<br>Gharagheizi et al. (2010) | X<br>Q<br>Q | 237<br><br>246 |
| 4,4-dimethyl-1-hexene<br>C$_8$H$_{16}$<br>[1647-08-1]<br>SUJVAMIXNUAJEY-UHFFFAOYSA-N | $1.2\times10^{-5}$<br>$1.6\times10^{-5}$<br>$1.3\times10^{-5}$ | | Yaws (2003)<br>Gharagheizi et al. (2012)<br>Gharagheizi et al. (2010) | X<br>Q<br>Q | 237<br><br>246 |
| 4,4-dimethyl-*cis*-2-hexene<br>C$_8$H$_{16}$<br>OQEVAISXHCRQGF-ALCCZGGFSA-N | $1.3\times10^{-5}$<br>$1.1\times10^{-5}$<br>$1.2\times10^{-5}$ | | Yaws (2003)<br>Gharagheizi et al. (2012)<br>Gharagheizi et al. (2010) | X<br>Q<br>Q | 237<br><br>246 |
| 4,4-dimethyl-*trans*-2-hexene<br>C$_8$H$_{16}$<br>[19550-83-5]<br>OQEVAISXHCRQGF-FNORWQNLSA-N | $1.3\times10^{-5}$<br>$1.1\times10^{-5}$<br>$1.2\times10^{-5}$ | | Yaws (2003)<br>Gharagheizi et al. (2012)<br>Gharagheizi et al. (2010) | X<br>Q<br>Q | 237<br><br>246 |
| 4,5-dimethyl-1-hexene<br>C$_8$H$_{16}$<br>[16106-59-5]<br>UFWIBUBEFUNVNI-UHFFFAOYSA-N | $1.2\times10^{-5}$<br>$1.7\times10^{-5}$<br>$1.3\times10^{-5}$ | | Yaws (2003)<br>Gharagheizi et al. (2012)<br>Gharagheizi et al. (2010) | X<br>Q<br>Q | 237<br><br>246 |
| 4,5-dimethyl-*cis*-2-hexene<br>C$_8$H$_{16}$<br>[65036-71-7]<br>OAVNNZUEVHDCKP-WAYWQWQTSA-N | $1.2\times10^{-5}$<br>$1.2\times10^{-5}$<br>$1.3\times10^{-5}$ | | Yaws (2003)<br>Gharagheizi et al. (2012)<br>Gharagheizi et al. (2010) | X<br>Q<br>Q | 237<br><br>246 |




Table A2.3: Aliphatic alkenes and cycloalkenes (...continued)

| Substance Formula (Trivial Name) [CAS Registry Number] InChIKey | $H_s^{cp}$ (at $T^{\ominus}$) $\left[\dfrac{\text{mol}}{\text{m}^3\,\text{Pa}}\right]$ | $\dfrac{\text{d}\ln H_s^{cp}}{\text{d}(1/T)}$ [K] | Reference | Type | Note |
|---|---|---|---|---|---|
| 4,5-dimethyl-*trans*-2-hexene C$_8$H$_{16}$ [66225-14-7] OAVNNZUEVHDCKP-AATRIKPKSA-N | $1.2\times10^{-5}$ $1.2\times10^{-5}$ $1.3\times10^{-5}$ | | Yaws (2003) Gharagheizi et al. (2012) Gharagheizi et al. (2010) | X Q Q | 237 246 |
| 5,5-dimethyl-1-hexene C$_8$H$_{16}$ [7116-86-1] KZJIOVQKSAOPOP-UHFFFAOYSA-N | $1.4\times10^{-5}$ $1.4\times10^{-5}$ $1.3\times10^{-5}$ | | Yaws (2003) Gharagheizi et al. (2012) Gharagheizi et al. (2010) | X Q Q | 237 246 |
| 5,5-dimethyl-*cis*-2-hexene C$_8$H$_{16}$ [39761-61-0] NWZJLSKAFZXSQH-WAYWQWQTSA-N | $1.2\times10^{-5}$ $1.1\times10^{-5}$ $1.2\times10^{-5}$ | | Yaws (2003) Gharagheizi et al. (2012) Gharagheizi et al. (2010) | X Q Q | 237 246 |
| 5,5-dimethyl-*trans*-2-hexene C$_8$H$_{16}$ [39782-43-9] NWZJLSKAFZXSQH-AATRIKPKSA-N | $1.3\times10^{-5}$ $1.0\times10^{-5}$ $1.2\times10^{-5}$ | | Yaws (2003) Gharagheizi et al. (2012) Gharagheizi et al. (2010) | X Q Q | 237 246 |
| 2-ethyl-1-hexene C$_8$H$_{16}$ [1632-16-2] XTVRLCUJHGUXCP-UHFFFAOYSA-N | $9.2\times10^{-6}$ $2.0\times10^{-5}$ $9.8\times10^{-6}$ $9.4\times10^{-6}$ | | Yaws (2003) Gharagheizi et al. (2012) Gharagheizi et al. (2010) Yaws (1999) | X Q Q ? | 237 246 21 |
| 3-ethyl-1-hexene C$_8$H$_{16}$ [3404-58-8] OLGHJTHQWQKJQQ-UHFFFAOYSA-N | $1.2\times10^{-5}$ $1.7\times10^{-5}$ $1.2\times10^{-5}$ | | Yaws (2003) Gharagheizi et al. (2012) Gharagheizi et al. (2010) | X Q Q | 237 246 |
| 3-ethyl-3-hexene C$_8$H$_{16}$ [16789-51-8] AUJLDZJNMXNESO-UHFFFAOYSA-N | $1.0\times10^{-5}$ $9.6\times10^{-6}$ $9.1\times10^{-6}$ | | Yaws (2003) Gharagheizi et al. (2012) Gharagheizi et al. (2010) | X Q Q | 237 246 |
| 3-ethyl-*cis*-2-hexene C$_8$H$_{16}$ [36880-72-5] QEMJIDSLEPYRLM-YVMONPNESA-N | $9.2\times10^{-6}$ $1.1\times10^{-5}$ $9.1\times10^{-6}$ | | Yaws (2003) Gharagheizi et al. (2012) Gharagheizi et al. (2010) | X Q Q | 237 246 |
| 3-ethyl-*trans*-2-hexene C$_8$H$_{16}$ [66225-15-8] QEMJIDSLEPYRLM-VMPITWQZSA-N | $9.2\times10^{-6}$ $1.1\times10^{-5}$ $9.1\times10^{-6}$ | | Yaws (2003) Gharagheizi et al. (2012) Gharagheizi et al. (2010) | X Q Q | 237 246 |
| 4-ethyl-1-hexene C$_8$H$_{16}$ [16746-85-3] OPMUAJRVOWSBTP-UHFFFAOYSA-N | $1.3\times10^{-5}$ $1.7\times10^{-5}$ $1.2\times10^{-5}$ | | Yaws (2003) Gharagheizi et al. (2012) Gharagheizi et al. (2010) | X Q Q | 237 246 |



Table A2.3: Aliphatic alkenes and cycloalkenes (. . . continued)

| Substance<br>Formula<br>(Trivial Name)<br>[CAS Registry Number]<br>InChIKey | $H_s^{cp}$<br>(at $T^\ominus$)<br>$\left[\dfrac{\text{mol}}{\text{m}^3\,\text{Pa}}\right]$ | $\dfrac{\text{d}\ln H_s^{cp}}{\text{d}(1/T)}$<br><br>[K] | Reference | Type | Note |
|---|---|---|---|---|---|
| 4-ethyl-*cis*-2-hexene<br>C$_8$H$_{16}$<br>[54616-49-8]<br>STHONQMAWQLWLX-DAXSKMNVSA-N | $1.1\times10^{-5}$<br>$1.3\times10^{-5}$<br>$1.1\times10^{-5}$ | | Yaws (2003)<br>Gharagheizi et al. (2012)<br>Gharagheizi et al. (2010) | X<br>Q<br>Q | 237<br><br>246 |
| 4-ethyl-*trans*-2-hexene<br>C$_8$H$_{16}$<br>[19781-63-6]<br>STHONQMAWQLWLX-QPJJXVBHSA-N | $1.1\times10^{-5}$<br>$1.3\times10^{-5}$<br>$1.1\times10^{-5}$ | | Yaws (2003)<br>Gharagheizi et al. (2012)<br>Gharagheizi et al. (2010) | X<br>Q<br>Q | 237<br><br>246 |
| 2,3,3-trimethyl-1-pentene<br>C$_8$H$_{16}$<br>[560-23-6]<br>TUSBCMPNIOJUBX-UHFFFAOYSA-N | $1.2\times10^{-5}$<br>$1.2\times10^{-5}$<br>$1.2\times10^{-5}$ | | Yaws (2003)<br>Gharagheizi et al. (2012)<br>Gharagheizi et al. (2010) | X<br>Q<br>Q | 237<br><br>246 |
| 2,3,4-trimethyl-1-pentene<br>C$_8$H$_{16}$<br>[565-76-4]<br>FAWUHEYSSPPNSH-UHFFFAOYSA-N | $1.2\times10^{-5}$<br>$1.1\times10^{-5}$<br>$1.3\times10^{-5}$ | | Yaws (2003)<br>Gharagheizi et al. (2012)<br>Gharagheizi et al. (2010) | X<br>Q<br>Q | 237<br><br>246 |
| 2,3,4-trimethyl-2-pentene<br>C$_8$H$_{16}$<br>[565-77-5]<br>SZFRZEBLZFTODC-UHFFFAOYSA-N | $1.0\times10^{-5}$<br>$6.8\times10^{-6}$<br>$1.1\times10^{-5}$ | | Yaws (2003)<br>Gharagheizi et al. (2012)<br>Gharagheizi et al. (2010) | X<br>Q<br>Q | 237<br><br>246 |
| 2,4,4-trimethyl-1-pentene<br>C$_8$H$_{16}$<br>[107-39-1]<br>FXNDIJDIPNCZQJ-UHFFFAOYSA-N | $1.2\times10^{-5}$<br>$1.3\times10^{-5}$<br>$9.0\times10^{-6}$<br>$1.2\times10^{-5}$<br>$2.6\times10^{-6}$<br>$1.1\times10^{-5}$<br>$1.2\times10^{-5}$ | | Yaws (2003)<br>HSDB (2015)<br>Gharagheizi et al. (2012)<br>Gharagheizi et al. (2010)<br>Modarresi et al. (2005)<br>Yao et al. (2002)<br>Yaws (1999) | X<br>Q<br>Q<br>Q<br>Q<br>Q<br>? | 237<br>99<br><br>246<br>247<br>229<br>21 |
| 2,4,4-trimethyl-2-pentene<br>C$_8$H$_{16}$<br>[107-40-4]<br>LAAVYEUJEMRIGF-UHFFFAOYSA-N | $1.2\times10^{-5}$<br>$1.1\times10^{-5}$<br>$6.8\times10^{-6}$<br>$1.2\times10^{-5}$<br>$5.9\times10^{-6}$<br>$1.0\times10^{-5}$<br>$1.2\times10^{-5}$ | | Yaws (2003)<br>HSDB (2015)<br>Gharagheizi et al. (2012)<br>Gharagheizi et al. (2010)<br>Modarresi et al. (2005)<br>Yao et al. (2002)<br>Yaws (1999) | X<br>Q<br>Q<br>Q<br>Q<br>Q<br>? | 237<br>99<br><br>246<br>247<br>229<br>21 |
| 3,3,4-trimethyl-1-pentene<br>C$_8$H$_{16}$<br>[560-22-5]<br>LLFHCOGPDCJMKY-UHFFFAOYSA-N | $1.3\times10^{-5}$<br>$1.5\times10^{-5}$<br>$1.4\times10^{-5}$ | | Yaws (2003)<br>Gharagheizi et al. (2012)<br>Gharagheizi et al. (2010) | X<br>Q<br>Q | 237<br><br>246 |
| 3,4,4-trimethyl-1-pentene<br>C$_8$H$_{16}$<br>[564-03-4]<br>BOPVNOMJIDZBQB-UHFFFAOYSA-N | $1.3\times10^{-5}$<br>$1.5\times10^{-5}$<br>$1.4\times10^{-5}$ | | Yaws (2003)<br>Gharagheizi et al. (2012)<br>Gharagheizi et al. (2010) | X<br>Q<br>Q | 237<br><br>246 |





Table A2.3: Aliphatic alkenes and cycloalkenes (...continued)

| Substance<br>Formula<br>(Trivial Name)<br>[CAS Registry Number]<br>InChIKey | $H_s^{cp}$<br>(at $T^\ominus$)<br><br>$\left[\dfrac{\mathrm{mol}}{\mathrm{m^3\,Pa}}\right]$ | $\dfrac{\mathrm{d}\ln H_s^{cp}}{\mathrm{d}(1/T)}$<br><br>[K] | Reference | Type | Note |
|---|---|---|---|---|---|
| 3,4,4-trimethyl-2-pentene<br>$C_8H_{16}$<br>[598-96-9]<br>FZQMZRXKWHQJAG-UHFFFAOYSA-N | $1.1 \times 10^{-5}$ | | HSDB (2015) | Q | 99 |
| 3,4,4-trimethyl-*cis*-2-pentene<br>$C_8H_{16}$<br>[39761-64-3]<br>FZQMZRXKWHQJAG-SREVYHEPSA-N | $1.1 \times 10^{-5}$<br>$8.9 \times 10^{-6}$<br>$1.2 \times 10^{-5}$ | | Yaws (2003)<br>Gharagheizi et al. (2012)<br>Gharagheizi et al. (2010) | X<br>Q<br>Q | 237<br><br>246 |
| 3,4,4-trimethyl-*trans*-2-pentene<br>$C_8H_{16}$<br>[39761-57-4]<br>FZQMZRXKWHQJAG-VOTSOKGWSA-N | $1.1 \times 10^{-5}$<br>$8.9 \times 10^{-6}$<br>$1.2 \times 10^{-5}$ | | Yaws (2003)<br>Gharagheizi et al. (2012)<br>Gharagheizi et al. (2010) | X<br>Q<br>Q | 237<br><br>246 |
| 2-methyl-3-ethyl-1-pentene<br>$C_8H_{16}$<br>[19780-66-6]<br>HPHHYSWOBXEIRG-UHFFFAOYSA-N | $1.1 \times 10^{-5}$<br>$1.2 \times 10^{-5}$<br>$1.1 \times 10^{-5}$ | | Yaws (2003)<br>Gharagheizi et al. (2012)<br>Gharagheizi et al. (2010) | X<br>Q<br>Q | 237<br><br>246 |
| 2-methyl-3-ethyl-2-pentene<br>$C_8H_{16}$<br>[19780-67-7]<br>FQYUGAXHZSQHMU-UHFFFAOYSA-N | $1.1 \times 10^{-5}$<br>$6.8 \times 10^{-6}$<br>$9.0 \times 10^{-6}$ | | Yaws (2003)<br>Gharagheizi et al. (2012)<br>Gharagheizi et al. (2010) | X<br>Q<br>Q | 237<br><br>246 |
| 3-methyl-2-ethyl-1-pentene<br>$C_8H_{16}$<br>[3404-67-9]<br>YXLCVBVDFKWWRW-UHFFFAOYSA-N | $1.2 \times 10^{-5}$<br>$1.2 \times 10^{-5}$<br>$1.1 \times 10^{-5}$ | | Yaws (2003)<br>Gharagheizi et al. (2012)<br>Gharagheizi et al. (2010) | X<br>Q<br>Q | 237<br><br>246 |
| 3-methyl-3-ethyl-1-pentene<br>$C_8H_{16}$<br>[6196-60-7]<br>PHHHEKOJKDYRIN-UHFFFAOYSA-N | $1.1 \times 10^{-5}$<br>$1.9 \times 10^{-5}$<br>$1.3 \times 10^{-5}$ | | Yaws (2003)<br>Gharagheizi et al. (2012)<br>Gharagheizi et al. (2010) | X<br>Q<br>Q | 237<br><br>246 |
| 4-methyl-2-ethyl-1-pentene<br>$C_8H_{16}$<br>[3404-80-6]<br>TVBQWTDYXVGWJL-UHFFFAOYSA-N | $1.2 \times 10^{-5}$<br>$1.2 \times 10^{-5}$<br>$1.1 \times 10^{-5}$ | | Yaws (2003)<br>Gharagheizi et al. (2012)<br>Gharagheizi et al. (2010) | X<br>Q<br>Q | 237<br><br>246 |
| 4-methyl-3-ethyl-1-pentene<br>$C_8H_{16}$<br>[61847-80-1]<br>DTNALCAUPPLROB-UHFFFAOYSA-N | $1.2 \times 10^{-5}$<br>$1.6 \times 10^{-5}$<br>$1.3 \times 10^{-5}$ | | Yaws (2003)<br>Gharagheizi et al. (2012)<br>Gharagheizi et al. (2010) | X<br>Q<br>Q | 237<br><br>246 |
| 4-methyl-3-ethyl-*cis*-2-pentene<br>$C_8H_{16}$<br>[42067-48-1]<br>DSTFDBMUTNIZGD-YVMONPNESA-N | $1.0 \times 10^{-5}$<br>$9.8 \times 10^{-6}$<br>$1.1 \times 10^{-5}$ | | Yaws (2003)<br>Gharagheizi et al. (2012)<br>Gharagheizi et al. (2010) | X<br>Q<br>Q | 237<br><br>246 |



Table A2.3: Aliphatic alkenes and cycloalkenes (…continued)

| Substance<br>Formula<br>(Trivial Name)<br>[CAS Registry Number]<br>InChIKey | $H_s^{cp}$<br>(at $T^{\ominus}$)<br><br>$\left[\dfrac{\text{mol}}{\text{m}^3\,\text{Pa}}\right]$ | $\dfrac{\text{d}\ln H_s^{cp}}{\text{d}(1/T)}$<br><br>[K] | Reference | Type | Note |
|---|---|---|---|---|---|
| 4-methyl-3-ethyl-*trans*-2-pentene<br>$C_8H_{16}$<br>[42067-49-2]<br>DSTFDBMUTNIZGD-VMPITWQZSA-N | $1.1\times10^{-5}$<br>$9.3\times10^{-6}$<br>$1.1\times10^{-5}$ | | Yaws (2003)<br>Gharagheizi et al. (2012)<br>Gharagheizi et al. (2010) | X<br>Q<br>Q | 237<br><br>246 |
| 2-propyl-1-pentene<br>$C_8H_{16}$<br>[15918-08-8]<br>FYUUBXZYRPRIHC-UHFFFAOYSA-N | $1.0\times10^{-5}$<br>$1.5\times10^{-5}$<br>$9.8\times10^{-6}$ | | Yaws (2003)<br>Gharagheizi et al. (2012)<br>Gharagheizi et al. (2010) | X<br>Q<br>Q | 237<br><br>246 |
| 2-isopropyl-1-pentene<br>$C_8H_{16}$<br>[16746-02-4]<br>QOUCFWFZPKWYRE-UHFFFAOYSA-N | $1.1\times10^{-5}$<br>$1.3\times10^{-5}$<br>$1.1\times10^{-5}$ | | Yaws (2003)<br>Gharagheizi et al. (2012)<br>Gharagheizi et al. (2010) | X<br>Q<br>Q | 237<br><br>246 |
| 3,3-dimethyl-2-ethyl-1-butene<br>$C_8H_{16}$<br>[18231-53-3]<br>KQRPZENSXZIOQO-UHFFFAOYSA-N | $1.2\times10^{-5}$<br>$1.2\times10^{-5}$<br>$1.2\times10^{-5}$ | | Yaws (2003)<br>Gharagheizi et al. (2012)<br>Gharagheizi et al. (2010) | X<br>Q<br>Q | 237<br><br>246 |
| 3-methyl-2-isopropyl-1-butene<br>$C_8H_{16}$<br>[111823-35-9]<br>GIBPFTYFAOOKOV-UHFFFAOYSA-N | $1.3\times10^{-5}$<br>$1.0\times10^{-5}$<br>$1.3\times10^{-5}$ | | Yaws (2003)<br>Gharagheizi et al. (2012)<br>Gharagheizi et al. (2010) | X<br>Q<br>Q | 237<br><br>246 |
| 1-nonene<br>$C_9H_{18}$<br>[124-11-8]<br>JRZJOMJEPLMPRA-UHFFFAOYSA-N | $1.1\times10^{-5}$<br>$1.2\times10^{-5}$<br>$1.2\times10^{-5}$<br>$1.2\times10^{-5}$<br>$1.2\times10^{-5}$<br>$1.2\times10^{-5}$<br>$1.6\times10^{-3}$<br>$1.9\times10^{-5}$<br>$8.9\times10^{-6}$<br>$6.5\times10^{-6}$<br>$1.1\times10^{-5}$<br>$1.7\times10^{-5}$<br>$1.2\times10^{-5}$<br>$1.0\times10^{-5}$<br>$1.2\times10^{-5}$<br>$1.2\times10^{-5}$<br>$1.2\times10^{-5}$ | | Brockbank (2013)<br>Plyasunov and Shock (2000)<br>Duchowicz et al. (2020)<br>Mackay et al. (2006a)<br>Mackay et al. (1993)<br>Yaws (2003)<br>Duchowicz et al. (2020)<br>Gharagheizi et al. (2012)<br>Gharagheizi et al. (2010)<br>Hilal et al. (2008)<br>Yaffe et al. (2003)<br>Yao et al. (2002)<br>English and Carroll (2001)<br>Nirmalakhandan et al. (1997)<br>Yaws (1999)<br>Yaws and Yang (1992)<br>Abraham et al. (1990) | L<br>L<br>V<br>V<br>V<br>X<br>Q<br>Q<br>Q<br>Q<br>Q<br>Q<br>Q<br>Q<br>?<br>?<br>? | <br><br>186<br><br><br>237<br><br><br>246<br><br>248, 272<br>229<br>230, 231<br><br>21<br>21<br> |
| 2-methyl-1-octene<br>$C_9H_{18}$<br>[4588-18-5]<br>FBEDQPGLIKZGIN-UHFFFAOYSA-N | $6.2\times10^{-6}$<br>$1.3\times10^{-5}$<br>$6.9\times10^{-6}$ | | Yaws (2003)<br>Gharagheizi et al. (2012)<br>Gharagheizi et al. (2010) | X<br>Q<br>Q | 237<br><br>246 |



Table A2.3: Aliphatic alkenes and cycloalkenes (...continued)

| Substance Formula (Trivial Name) [CAS Registry Number] InChIKey | $H_s^{cp}$ (at $T^{\ominus}$) $\left[\dfrac{\mathrm{mol}}{\mathrm{m^3\,Pa}}\right]$ | $\dfrac{\mathrm{d}\ln H_s^{cp}}{\mathrm{d}(1/T)}$ [K] | Reference | Type | Note |
|---|---|---|---|---|---|
| 1-decene $C_{10}H_{20}$ [872-05-9] AFFLGGQVNFXPEV-UHFFFAOYSA-N | $8.6\times10^{-6}$ $1.8\times10^{-5}$ $3.7\times10^{-6}$ $3.3\times10^{-6}$ $3.7\times10^{-6}$ $1.6\times10^{-3}$ $2.4\times10^{-5}$ $7.3\times10^{-6}$ $4.2\times10^{-6}$ $4.7\times10^{-6}$ | | Brockbank (2013) Duchowicz et al. (2020) HSDB (2015) Mackay et al. (1993) Yaws (2003) Duchowicz et al. (2020) Gharagheizi et al. (2012) Gharagheizi et al. (2010) Hilal et al. (2008) Yaws (1999) | L V V V X Q Q Q Q ? | 186 237 246 21 |
| 2-methyl-1-nonene $C_{10}H_{20}$ [2980-71-4] YLZQHQUVNZVGOK-UHFFFAOYSA-N | $4.3\times10^{-6}$ $1.0\times10^{-5}$ $5.2\times10^{-6}$ | | Yaws (2003) Gharagheizi et al. (2012) Gharagheizi et al. (2010) | X Q Q | 237 246 |
| 1-undecene $C_{11}H_{22}$ [821-95-4] DCTOHCCUXLBQMS-UHFFFAOYSA-N | $4.5\times10^{-6}$ $6.7\times10^{-6}$ $2.9\times10^{-5}$ $6.7\times10^{-6}$ $2.2\times10^{-6}$ $1.2\times10^{-5}$ $4.6\times10^{-6}$ | | Yaws (2003) HSDB (2015) Gharagheizi et al. (2012) Gharagheizi et al. (2010) Hilal et al. (2008) Yao et al. (2002) Yaws (1999) | X Q Q Q Q Q ? | 237 99 246 229 21 |
| 2-methyl-1-decene $C_{11}H_{22}$ [13151-27-4] HLMACKQLXSEXIY-UHFFFAOYSA-N | $3.4\times10^{-6}$ $1.8\times10^{-5}$ $4.4\times10^{-6}$ | | Yaws (2003) Gharagheizi et al. (2012) Gharagheizi et al. (2010) | X Q Q | 237 246 |
| 1-dodecene $C_{12}H_{24}$ [112-41-4] CRSBERNSMYQZNG-UHFFFAOYSA-N | $5.1\times10^{-6}$ $2.3\times10^{-6}$ $3.4\times10^{-5}$ $7.1\times10^{-6}$ $1.5\times10^{-6}$ $9.9\times10^{-6}$ $5.2\times10^{-6}$ | | Yaws (2003) HSDB (2015) Gharagheizi et al. (2012) Gharagheizi et al. (2010) Hilal et al. (2008) Yao et al. (2002) Yaws (1999) | X Q Q Q Q Q ? | 237 99 246 229 21 |
| 2-methyl-1-undecene $C_{12}H_{24}$ [18516-37-5] SJVKHZYVCVKEGM-UHFFFAOYSA-N | $3.4\times10^{-6}$ $4.3\times10^{-6}$ | | Yaws (2003) Gharagheizi et al. (2010) | X Q | 237 246 |
| 2,2,4,6,6-pentamethyl-3-heptene $C_{12}H_{24}$ [123-48-8] NBUMCEJRJRRLCA-UHFFFAOYSA-N | $3.6\times10^{-6}$ $5.2\times10^{-7}$ $1.8\times10^{-5}$ $1.5\times10^{-5}$ | | Zhang et al. (2010) Zhang et al. (2010) Zhang et al. (2010) Zhang et al. (2010) | Q Q Q Q | 287, 288 287, 289 287, 290 287, 291 |



Table A2.3: Aliphatic alkenes and cycloalkenes (...continued)

| Substance Formula (Trivial Name) [CAS Registry Number] InChIKey | $H_s^{cp}$ (at $T^{\ominus}$) $\left[\dfrac{\text{mol}}{\text{m}^3\,\text{Pa}}\right]$ | $\dfrac{\text{d}\ln H_s^{cp}}{\text{d}(1/T)}$ [K] | Reference | Type | Note |
|---|---|---|---|---|---|
| 1-tridecene | $5.8\times10^{-6}$ | | Yaws (2003) | X | 258 |
| $C_{13}H_{26}$ | $5.8\times10^{-6}$ | | Yaws (2003) | X | 237 |
| [2437-56-1] | $9.4\times10^{-6}$ | | Dupeux et al. (2022) | Q | 259 |
| VQOXUMQBYILCKR-UHFFFAOYSA-N | $3.8\times10^{-6}$ | | HSDB (2015) | Q | 99 |
| | $3.9\times10^{-5}$ | | Gharagheizi et al. (2012) | Q | |
| | $8.8\times10^{-6}$ | | Gharagheizi et al. (2010) | Q | 246 |
| | $9.2\times10^{-6}$ | | Yao et al. (2002) | Q | 229 |
| | $5.9\times10^{-6}$ | | Yaws (1999) | ? | 21 |
| 2-methyl-1-dodecene | $4.1\times10^{-6}$ | | Yaws (2003) | X | 237 |
| $C_{13}H_{26}$ | $2.4\times10^{-5}$ | | Gharagheizi et al. (2012) | Q | |
| [16435-49-7] | $4.9\times10^{-6}$ | | Gharagheizi et al. (2010) | Q | 246 |
| PWRBDKMPAZFCSV-UHFFFAOYSA-N | | | | | |
| 1-tetradecene | $3.0\times10^{-6}$ | | Brockbank (2013) | L | |
| $C_{14}H_{28}$ | $1.4\times10^{-5}$ | | Yaws (2003) | X | 237 |
| [1120-36-1] | $1.2\times10^{-6}$ | | HSDB (2015) | Q | 99 |
| HFDVRLIODXPAHB-UHFFFAOYSA-N | $4.3\times10^{-5}$ | | Gharagheizi et al. (2012) | Q | |
| | $1.3\times10^{-5}$ | | Gharagheizi et al. (2010) | Q | 246 |
| | $1.5\times10^{-5}$ | | Yaws (1999) | ? | 21 |
| 2-tetradecene | $7.5\times10^{-6}$ | | Yaws (2003) | X | 237 |
| $C_{14}H_{28}$ | $4.0\times10^{-5}$ | | Gharagheizi et al. (2012) | Q | |
| [638-60-8] | $4.6\times10^{-6}$ | | Gharagheizi et al. (2010) | Q | 246 |
| OBDUMNZXAIUUTH-UHFFFAOYSA-N | | | | | |
| 2-methyl-1-tridecene | $8.9\times10^{-6}$ | | Yaws (2003) | X | 237 |
| $C_{14}H_{28}$ | $3.0\times10^{-5}$ | | Gharagheizi et al. (2012) | Q | |
| [18094-01-4] | $6.9\times10^{-6}$ | | Gharagheizi et al. (2010) | Q | 246 |
| VNBHQOHLCULRDN-UHFFFAOYSA-N | | | | | |
| 1-pentadecene | $2.2\times10^{-6}$ | | Brockbank (2013) | L | |
| $C_{15}H_{30}$ | $3.5\times10^{-5}$ | | Yaws (2003) | X | 237 |
| [13360-61-7] | $4.7\times10^{-5}$ | | Gharagheizi et al. (2012) | Q | |
| PJLHTVIBELQURV-UHFFFAOYSA-N | $2.6\times10^{-5}$ | | Gharagheizi et al. (2010) | Q | 246 |
| | $8.7\times10^{-5}$ | | Yao et al. (2002) | Q | 229, 267 |
| | $3.6\times10^{-5}$ | | Yaws (1999) | ? | 21 |
| 2-methyl-1-tetradecene | $1.1\times10^{-5}$ | | Yaws (2003) | X | 237 |
| $C_{15}H_{30}$ | $2.9\times10^{-5}$ | | Gharagheizi et al. (2012) | Q | |
| [52254-38-3] | $1.2\times10^{-5}$ | | Gharagheizi et al. (2010) | Q | 246 |
| WSNMNSLVXDWAFZ-UHFFFAOYSA-N | | | | | |
| 1-hexadecene | $5.7\times10^{-5}$ | | Yaws (2003) | X | 237 |
| $C_{16}H_{32}$ | $4.9\times10^{-5}$ | | Gharagheizi et al. (2012) | Q | |
| [629-73-2] | $6.7\times10^{-5}$ | | Gharagheizi et al. (2010) | Q | 246 |
| GQEZCXVZFLOKMC-UHFFFAOYSA-N | $5.8\times10^{-5}$ | | Yaws (1999) | ? | 21 |





Table A2.3: Aliphatic alkenes and cycloalkenes (...continued)

| Substance Formula (Trivial Name) [CAS Registry Number] InChIKey | $H_s^{cp}$ (at $T^{\ominus}$) $\left[\dfrac{\text{mol}}{\text{m}^3\,\text{Pa}}\right]$ | $\dfrac{\text{d}\ln H_s^{cp}}{\text{d}(1/T)}$ [K] | Reference | Type | Note |
|---|---|---|---|---|---|
| 2-methyl-1-pentadecene $C_{16}H_{32}$ [29833-69-0] FWQJRKLMXMTXDY-UHFFFAOYSA-N | $2.5\times10^{-5}$ $3.1\times10^{-5}$ $2.8\times10^{-5}$ | | Yaws (2003) Gharagheizi et al. (2012) Gharagheizi et al. (2010) | X Q Q | 237 246 |
| 2,4-dimethylpentadecane $C_{17}H_{36}$ [61868-07-3] TXGHMPJBDYZJLF-UHFFFAOYSA-N | $1.1\times10^{-5}$ $4.0\times10^{-6}$ $2.1\times10^{-5}$ | | Yaws (2003) Gharagheizi et al. (2012) Gharagheizi et al. (2010) | X Q Q | 237 246 |
| 1,2-butadiene $C_4H_6$ [590-19-2] QNRMTGGDHLBXQZ-UHFFFAOYSA-N | $1.2\times10^{-4}$ $1.0\times10^{-4}$ $1.5\times10^{-4}$ $1.2\times10^{-4}$ $1.1\times10^{-4}$ $1.2\times10^{-4}$ | | Yaws (2003) HSDB (2015) Gharagheizi et al. (2012) Gharagheizi et al. (2010) Hilal et al. (2008) Yaws (1999) | X Q Q Q Q ? | 237 99 246 21 |
| 1,3-butadiene $C_4H_6$ [106-99-0] KAKZBPTYRLMSJV-UHFFFAOYSA-N | $1.4\times10^{-4}$ $1.3\times10^{-4}$ $1.4\times10^{-4}$ $1.4\times10^{-4}$ $1.3\times10^{-4}$ $1.3\times10^{-4}$ $1.3\times10^{-4}$ $3.9\times10^{-6}$ $4.8\times10^{-5}$ $5.0\times10^{-5}$ $1.6\times10^{-4}$ $1.2\times10^{-4}$ $1.4\times10^{-4}$ $1.5\times10^{-4}$ $1.5\times10^{-4}$ $5.6\times10^{-3}$ $6.3\times10^{-4}$ $1.4\times10^{-4}$ $2.0\times10^{-4}$ $2.7\times10^{-4}$ $2.0\times10^{-4}$ $9.9\times10^{-5}$ $1.2\times10^{-4}$ $1.4\times10^{-4}$ $1.8\times10^{-4}$ $1.8\times10^{-4}$  $1.3\times10^{-5}$ $1.3\times10^{-4}$ $2.5\times10^{-4}$ $1.3\times10^{-4}$ $1.8\times10^{-4}$ | 3200  4500                     3600 | Plyasunov and Shock (2000) Mackay and Shiu (1981) Wilhelm et al. (1977) Ross and Hudson (1957) Duchowicz et al. (2020) HSDB (2015) Mackay et al. (2006a) Lide and Frederikse (1995) Mackay et al. (1993) Hwang et al. (1992) Hine and Mookerjee (1975) Irmann (1965) Yaws (2003) Irmann (1965) Hayer et al. (2022) Duchowicz et al. (2020) Wang et al. (2017) Wang et al. (2017) Wang et al. (2017) Gharagheizi et al. (2012) Raventos-Duran et al. (2010) Raventos-Duran et al. (2010) Raventos-Duran et al. (2010) Gharagheizi et al. (2010) Hilal et al. (2008) Modarresi et al. (2007) Kühne et al. (2005) Modarresi et al. (2005) Yaffe et al. (2003) Yao et al. (2002) English and Carroll (2001) Suzuki et al. (1992) | L L L M V V V V V V V V X C Q Q Q Q Q Q Q Q Q Q Q Q Q Q Q Q Q Q | 186 237 20 80, 238 80, 239 80, 240 242, 243 244 245 246 67 247 248, 249 229 230, 274 232 |



Table A2.3: Aliphatic alkenes and cycloalkenes (. . . continued)

| Substance<br>Formula<br>(Trivial Name)<br>[CAS Registry Number]<br>InChIKey | $H_s^{cp}$<br>(at $T^{\ominus}$)<br>$\left[\dfrac{\text{mol}}{\text{m}^3\,\text{Pa}}\right]$ | $\dfrac{\text{d}\ln H_s^{cp}}{\text{d}(1/T)}$<br><br>[K] | Reference | Type | Note |
|---|---|---|---|---|---|
| | $9.2\times10^{-5}$ | | Nirmalakhandan and Speece (1988) | Q | |
| | | 4100 | Kühne et al. (2005) | ? | |
| | $1.4\times10^{-4}$ | | Yaws (1999) | ? | 21 |
| | $1.4\times10^{-4}$ | | Yaws and Yang (1992) | ? | 21 |
| 3-methyl-1,2-butadiene | $1.1\times10^{-4}$ | | Yaws (2003) | X | 237 |
| $C_5H_8$ | $7.2\times10^{-5}$ | | Gharagheizi et al. (2012) | Q | |
| (2-methyl-2,3-butadiene) | $9.5\times10^{-5}$ | | Gharagheizi et al. (2010) | Q | 246 |
| [598-25-4] | $1.0\times10^{-5}$ | | Modarresi et al. (2005) | Q | 247 |
| PAKGDPSCXSUALC-UHFFFAOYSA-N | $1.1\times10^{-4}$ | | Yaws (1999) | ? | 21 |
| 2-methyl-1,3-butadiene | $1.3\times10^{-4}$ | 2700 | Plyasunov and Shock (2000) | L | |
| $C_5H_8$ | $1.3\times10^{-4}$ | | Mackay and Shiu (1981) | L | |
| (isoprene) | $3.0\times10^{-4}$ | | Schuhfried et al. (2015) | M | |
| [78-79-5] | $3.4\times10^{-4}$ | 4400 | Leng et al. (2013) | M | |
| RRHGJUQNOFWUDK-UHFFFAOYSA-N | $1.2\times10^{-4}$ | 4400 | Ooki and Yokouchi (2011) | M | 70 |
| | $2.9\times10^{-4}$ | | Karl et al. (2003) | M | 28 |
| | $1.3\times10^{-4}$ | | Duchowicz et al. (2020) | V | 186 |
| | $1.3\times10^{-4}$ | | Martins et al. (2017) | V | 315 |
| | $1.3\times10^{-4}$ | | HSDB (2015) | V | |
| | $1.3\times10^{-4}$ | | Mackay et al. (2006a) | V | |
| | $1.3\times10^{-4}$ | | Copolovici and Niinemets (2005) | V | |
| | $1.3\times10^{-4}$ | | Mackay et al. (1993) | V | |
| | $1.3\times10^{-4}$ | | Hine and Mookerjee (1975) | V | |
| | $1.3\times10^{-4}$ | | Yaws (2003) | X | 237 |
| | $1.8\times10^{-3}$ | | Duchowicz et al. (2020) | Q | |
| | $4.1\times10^{-4}$ | | Wang et al. (2017) | Q | 80, 238 |
| | $1.6\times10^{-4}$ | | Wang et al. (2017) | Q | 80, 239 |
| | $1.9\times10^{-4}$ | | Wang et al. (2017) | Q | 80, 240 |
| | $1.8\times10^{-4}$ | | Gharagheizi et al. (2012) | Q | |
| | $1.6\times10^{-4}$ | | Raventos-Duran et al. (2010) | Q | 242, 243 |
| | $1.6\times10^{-4}$ | | Raventos-Duran et al. (2010) | Q | 244 |
| | $7.8\times10^{-5}$ | | Raventos-Duran et al. (2010) | Q | 245 |
| | $1.0\times10^{-4}$ | | Gharagheizi et al. (2010) | Q | 246 |
| | $2.7\times10^{-4}$ | | Hilal et al. (2008) | Q | |
| | $1.1\times10^{-5}$ | | Modarresi et al. (2005) | Q | 247 |
| | $1.3\times10^{-4}$ | | Yaffe et al. (2003) | Q | 248, 249 |
| | $1.0\times10^{-4}$ | | Yao et al. (2002) | Q | 229 |
| | $1.1\times10^{-4}$ | | Yao et al. (2002) | Q | 229 |
| | $1.0\times10^{-4}$ | | English and Carroll (2001) | Q | 230, 231 |
| | $1.4\times10^{-4}$ | | Suzuki et al. (1992) | Q | 232 |
| | $6.7\times10^{-5}$ | | Nirmalakhandan and Speece (1988) | Q | |
| | $1.1\times10^{-4}$ | | Yaws (1999) | ? | 21 |
| | $1.3\times10^{-4}$ | | Yaws (1999) | ? | 21 |
| | $1.3\times10^{-4}$ | | Yaws and Yang (1992) | ? | 21 |



Table A2.3: Aliphatic alkenes and cycloalkenes (... continued)

| Substance Formula (Trivial Name) [CAS Registry Number] InChIKey | $H_s^{cp}$ (at $T^{\ominus}$) $\left[\dfrac{\mathrm{mol}}{\mathrm{m^3\,Pa}}\right]$ | $\dfrac{\mathrm{d}\ln H_s^{cp}}{\mathrm{d}(1/T)}$ [K] | Reference | Type | Note |
|---|---|---|---|---|---|
| 1,2-pentadiene $C_5H_8$ [591-95-7] LVMTVPFRTKXRPH-UHFFFAOYSA-N | $1.1\times10^{-4}$ $1.2\times10^{-4}$ $1.1\times10^{-4}$ $9.7\times10^{-5}$ $1.1\times10^{-4}$ | | Yaws (2003) Gharagheizi et al. (2012) Gharagheizi et al. (2010) Hilal et al. (2008) Yaws (1999) | X Q Q Q ? | 237 246 21 |
| 1,3-pentadiene $C_5H_8$ [504-60-9] PMJHHCWVYXUKFD-UHFFFAOYSA-N | $1.4\times10^{-4}$ | | HSDB (2015) | Q | 99 |
| *trans*-1,3-pentadiene $C_5H_8$ [2004-70-8] PMJHHCWVYXUKFD-SNAWJCMRSA-N | $1.0\times10^{-4}$ $8.2\times10^{-5}$ $2.2\times10^{-4}$ $1.0\times10^{-4}$ $1.1\times10^{-4}$ | | Yaws (2003) HSDB (2015) Gharagheizi et al. (2012) Gharagheizi et al. (2010) Yaws (1999) | X Q Q Q ? | 237 99 246 21 |
| *cis*-1,3-pentadiene $C_5H_8$ [1574-41-0] PMJHHCWVYXUKFD-PLNGDYQASA-N | $1.1\times10^{-4}$ $2.4\times10^{-4}$ $1.0\times10^{-4}$ $1.1\times10^{-4}$ | | Yaws (2003) Gharagheizi et al. (2012) Gharagheizi et al. (2010) Yaws (1999) | X Q Q ? | 237 246 21 |
| 1,4-pentadiene $C_5H_8$ [591-93-5] QYZLKGVUSQXAMU-UHFFFAOYSA-N | $8.0\times10^{-5}$ $8.3\times10^{-5}$ $8.2\times10^{-5}$ $8.4\times10^{-5}$ $8.4\times10^{-5}$ $8.2\times10^{-5}$ $8.3\times10^{-5}$ $5.6\times10^{-3}$ $2.1\times10^{-4}$ $1.6\times10^{-4}$ $7.8\times10^{-5}$ $7.8\times10^{-5}$ $8.5\times10^{-5}$ $9.9\times10^{-5}$ $8.2\times10^{-5}$ $1.2\times10^{-5}$ $8.4\times10^{-5}$ $1.1\times10^{-4}$ $9.9\times10^{-5}$ $1.5\times10^{-4}$ $1.6\times10^{-4}$ $7.3\times10^{-5}$ $8.4\times10^{-5}$ $8.3\times10^{-5}$ | | Plyasunov and Shock (2000) Mackay and Shiu (1981) Duchowicz et al. (2020) Mackay et al. (2006a) Mackay et al. (1993) Hine and Mookerjee (1975) Yaws (2003) Duchowicz et al. (2020) Gharagheizi et al. (2012) Raventos-Duran et al. (2010) Raventos-Duran et al. (2010) Raventos-Duran et al. (2010) Gharagheizi et al. (2010) Hilal et al. (2008) Modarresi et al. (2007) Modarresi et al. (2005) Yaffe et al. (2003) Yao et al. (2002) English and Carroll (2001) Russell et al. (1992) Suzuki et al. (1992) Nirmalakhandan and Speece (1988) Yaws (1999) Yaws and Yang (1992) | L L V V V V X Q Q Q Q Q Q Q Q Q Q Q Q Q Q Q ? ? | 186 237 242, 243 244 245 246 67 247 248, 249 229, 267 230, 260 279 232 21 21 |



Table A2.3: Aliphatic alkenes and cycloalkenes (...continued)

| Substance<br>Formula<br>(Trivial Name)<br>[CAS Registry Number]<br>InChIKey | $H_s^{cp}$<br>(at $T^\ominus$)<br>$\left[\dfrac{\text{mol}}{\text{m}^3\,\text{Pa}}\right]$ | $\dfrac{\text{d}\ln H_s^{cp}}{\text{d}(1/T)}$<br><br>[K] | Reference | Type | Note |
|---|---|---|---|---|---|
| 2,3-pentadiene<br>$C_5H_8$<br>[591-96-8]<br>PODAMDNJNMAKAZ-UHFFFAOYSA-N | $1.1\times10^{-4}$<br>$8.8\times10^{-5}$<br>$9.6\times10^{-5}$<br>$1.1\times10^{-4}$<br>$1.1\times10^{-4}$ | | Yaws (2003)<br>Gharagheizi et al. (2012)<br>Gharagheizi et al. (2010)<br>Hilal et al. (2008)<br>Yaws (1999) | X<br>Q<br>Q<br>Q<br>? | 237<br><br>246<br><br>21 |
| 1,2-hexadiene<br>$C_6H_{10}$<br>[592-44-9]<br>XIAJQOBRHVKGSP-UHFFFAOYSA-N | $7.5\times10^{-5}$<br>$1.2\times10^{-4}$<br>$9.8\times10^{-5}$ | | Yaws (2003)<br>Gharagheizi et al. (2012)<br>Gharagheizi et al. (2010) | X<br>Q<br>Q | 237<br><br>246 |
| cis-1,3-hexadiene<br>$C_6H_{10}$<br>[14596-92-0]<br>AHAREKHAZNPPMI-WAYWQWQTSA-N | $7.7\times10^{-5}$<br>$2.1\times10^{-4}$<br>$8.5\times10^{-5}$ | | Yaws (2003)<br>Gharagheizi et al. (2012)<br>Gharagheizi et al. (2010) | X<br>Q<br>Q | 237<br><br>246 |
| trans-1,3-hexadiene<br>$C_6H_{10}$<br>[20237-34-7]<br>AHAREKHAZNPPMI-AATRIKPKSA-N | $7.7\times10^{-5}$<br>$2.1\times10^{-4}$<br>$8.5\times10^{-5}$ | | Yaws (2003)<br>Gharagheizi et al. (2012)<br>Gharagheizi et al. (2010) | X<br>Q<br>Q | 237<br><br>246 |
| 1,4-hexadiene<br>$C_6H_{10}$<br>[592-45-0]<br>PRBHEGAFLDMLAL-UHFFFAOYSA-N | $8.4\times10^{-5}$ | | HSDB (2015) | Q | 99 |
| cis-1,4-hexadiene<br>$C_6H_{10}$<br>[7318-67-4]<br>PRBHEGAFLDMLAL-XQRVVYSFSA-N | $8.3\times10^{-5}$<br>$1.7\times10^{-4}$<br>$8.7\times10^{-5}$ | | Yaws (2003)<br>Gharagheizi et al. (2012)<br>Gharagheizi et al. (2010) | X<br>Q<br>Q | 237<br><br>246 |
| trans-1,4-hexadiene<br>$C_6H_{10}$<br>[7319-00-8]<br>PRBHEGAFLDMLAL-GQCTYLIASA-N | $8.3\times10^{-5}$<br>$1.7\times10^{-4}$<br>$8.7\times10^{-5}$ | | Yaws (2003)<br>Gharagheizi et al. (2012)<br>Gharagheizi et al. (2010) | X<br>Q<br>Q | 237<br><br>246 |
| 1,5-hexadiene<br>$C_6H_{10}$<br>[592-42-7]<br>PYGSKMBEVAICCR-UHFFFAOYSA-N | $6.8\times10^{-5}$<br>$7.0\times10^{-5}$<br>$6.9\times10^{-5}$<br>$6.7\times10^{-5}$<br>$7.3\times10^{-5}$<br>$8.1\times10^{-5}$<br>$5.6\times10^{-3}$<br>$2.0\times10^{-4}$<br>$7.7\times10^{-5}$<br>$5.8\times10^{-5}$<br>$1.0\times10^{-4}$<br>$9.4\times10^{-6}$<br>$5.8\times10^{-5}$<br>$3.5\times10^{-5}$<br>$7.5\times10^{-5}$ | | Plyasunov and Shock (2000)<br>Duchowicz et al. (2020)<br>Mackay et al. (2006a)<br>Hwang et al. (1992)<br>Hine and Mookerjee (1975)<br>Yaws (2003)<br>Duchowicz et al. (2020)<br>Gharagheizi et al. (2012)<br>Gharagheizi et al. (2010)<br>Hilal et al. (2008)<br>Modarresi et al. (2007)<br>Modarresi et al. (2005)<br>Yaffe et al. (2003)<br>Yao et al. (2002)<br>English and Carroll (2001) | L<br>V<br>V<br>V<br>V<br>X<br>Q<br>Q<br>Q<br>Q<br>Q<br>Q<br>Q<br>Q<br>Q | <br>186<br><br><br><br>237<br><br><br>246<br><br>67<br>247<br>248, 249<br>229<br>230, 231 |



Table A2.3: Aliphatic alkenes and cycloalkenes (. . . continued)

| Substance Formula (Trivial Name) [CAS Registry Number] InChIKey | $H_s^{cp}$ (at $T^\ominus$) $\left[\dfrac{\text{mol}}{\text{m}^3\,\text{Pa}}\right]$ | $\dfrac{\text{d}\ln H_s^{cp}}{\text{d}(1/T)}$ [K] | Reference | Type | Note |
|---|---|---|---|---|---|
| | $1.2\times10^{-4}$ | | Suzuki et al. (1992) | Q | 232 |
| | $5.8\times10^{-5}$ | | Nirmalakhandan and Speece (1988) | Q | |
| | $8.4\times10^{-5}$ | | Yaws (1999) | ? | 21 |
| 2,3-hexadiene $C_6H_{10}$ [592-49-4] DPUXQWOMYBMHRN-UHFFFAOYSA-N | $8.1\times10^{-5}$ $6.7\times10^{-5}$ $8.0\times10^{-5}$ | | Yaws (2003) Gharagheizi et al. (2012) Gharagheizi et al. (2010) | X Q Q | 237 246 |
| *cis*-2,*cis*-4-hexadiene $C_6H_{10}$ [6108-61-8] APPOKADJQUIAHP-GLIMQPGKSA-N | $7.1\times10^{-5}$ $1.8\times10^{-4}$ $7.5\times10^{-5}$ | | Yaws (2003) Gharagheizi et al. (2012) Gharagheizi et al. (2010) | X Q Q | 237 246 |
| *cis*-2,*trans*-4-hexadiene $C_6H_{10}$ [5194-50-3] APPOKADJQUIAHP-CIIODKQPSA-N | $6.8\times10^{-5}$ $1.8\times10^{-4}$ $7.5\times10^{-5}$ $7.0\times10^{-5}$ | | Yaws (2003) Gharagheizi et al. (2012) Gharagheizi et al. (2010) Yaws (1999) | X Q Q ? | 237 246 21 |
| *trans*-2,*trans*-4-hexadiene $C_6H_{10}$ [5194-51-4] APPOKADJQUIAHP-GGWOSOGESA-N | $7.0\times10^{-5}$ $1.7\times10^{-4}$ $7.5\times10^{-5}$ $7.2\times10^{-5}$ | | Yaws (2003) Gharagheizi et al. (2012) Gharagheizi et al. (2010) Yaws (1999) | X Q Q ? | 237 246 21 |
| 2-methyl-1,4-pentadiene $C_6H_{10}$ [763-30-4] DRWYRROCDFQZQF-UHFFFAOYSA-N | $8.8\times10^{-5}$ $1.1\times10^{-4}$ $9.8\times10^{-5}$ | | Yaws (2003) Gharagheizi et al. (2012) Gharagheizi et al. (2010) | X Q Q | 237 246 |
| 2-methyl-1,*cis*-3-pentadiene $C_6H_{10}$ [1501-60-6] RCJMVGJKROQDCB-PLNGDYQASA-N | $7.3\times10^{-5}$ $1.6\times10^{-4}$ $7.5\times10^{-5}$ | | Yaws (2003) Gharagheizi et al. (2012) Gharagheizi et al. (2010) | X Q Q | 237 246 |
| 2-methyl-1,*trans*-3-pentadiene $C_6H_{10}$ [926-54-5] RCJMVGJKROQDCB-SNAWJCMRSA-N | $7.3\times10^{-5}$ $1.6\times10^{-4}$ $7.5\times10^{-5}$ | | Yaws (2003) Gharagheizi et al. (2012) Gharagheizi et al. (2010) | X Q Q | 237 246 |
| 2-methyl-2,3-pentadiene $C_6H_{10}$ [3043-33-2] JWMDOGMKTRMFDS-UHFFFAOYSA-N | $7.6\times10^{-5}$ $5.2\times10^{-5}$ $8.2\times10^{-5}$ | | Yaws (2003) Gharagheizi et al. (2012) Gharagheizi et al. (2010) | X Q Q | 237 246 |
| 3-methyl-1,2-pentadiene $C_6H_{10}$ [7417-48-3] INFFCVIZNSUFGK-UHFFFAOYSA-N | $7.8\times10^{-5}$ $7.0\times10^{-5}$ $7.9\times10^{-5}$ | | Yaws (2003) Gharagheizi et al. (2012) Gharagheizi et al. (2010) | X Q Q | 237 246 |





Table A2.3: Aliphatic alkenes and cycloalkenes (...continued)

| Substance Formula (Trivial Name) [CAS Registry Number] InChIKey | $H_s^{cp}$ (at $T^{\ominus}$) $\left[\dfrac{\mathrm{mol}}{\mathrm{m^3\,Pa}}\right]$ | $\dfrac{\mathrm{d}\ln H_s^{cp}}{\mathrm{d}(1/T)}$ [K] | Reference | Type | Note |
|---|---|---|---|---|---|
| 3-methyl-1,*cis*-3-pentadiene $C_6H_{10}$ [2787-45-3] BOGRNZQRTNVZCZ-WAYWQWQTSA-N | $7.2\times10^{-5}$ $1.6\times10^{-4}$ $7.5\times10^{-5}$ | | Yaws (2003) Gharagheizi et al. (2012) Gharagheizi et al. (2010) | X Q Q | 237 246 |
| 3-methyl-1,*trans*-3-pentadiene $C_6H_{10}$ [2787-43-1] BOGRNZQRTNVZCZ-AATRIKPKSA-N | $7.2\times10^{-5}$ $1.6\times10^{-4}$ $7.5\times10^{-5}$ | | Yaws (2003) Gharagheizi et al. (2012) Gharagheizi et al. (2010) | X Q Q | 237 246 |
| 3-methyl-1,4-pentadiene $C_6H_{10}$ [1115-08-8] IKQUUYYDRTYXAP-UHFFFAOYSA-N | $8.6\times10^{-5}$ $1.9\times10^{-4}$ $9.6\times10^{-5}$ | | Yaws (2003) Gharagheizi et al. (2012) Gharagheizi et al. (2010) | X Q Q | 237 246 |
| 4-methyl-1,2-pentadiene $C_6H_{10}$ [13643-05-5] CAAAXQFHDYHTTC-UHFFFAOYSA-N | $7.8\times10^{-5}$ $9.7\times10^{-5}$ $9.2\times10^{-5}$ | | Yaws (2003) Gharagheizi et al. (2012) Gharagheizi et al. (2010) | X Q Q | 237 246 |
| 4-methyl-1,3-pentadiene $C_6H_{10}$ [926-56-7] CJSBUWDGPXGFGA-UHFFFAOYSA-N | $7.3\times10^{-5}$ $1.6\times10^{-4}$ $7.5\times10^{-5}$ | | Yaws (2003) Gharagheizi et al. (2012) Gharagheizi et al. (2010) | X Q Q | 237 246 |
| 2-ethyl-1,3-butadiene $C_6H_{10}$ [3404-63-5] IGLWCQMNTGCUBB-UHFFFAOYSA-N | $7.4\times10^{-5}$ $2.2\times10^{-4}$ $8.3\times10^{-5}$ | | Yaws (2003) Gharagheizi et al. (2012) Gharagheizi et al. (2010) | X Q Q | 237 246 |
| 2,3-dimethyl-1,3-butadiene $C_6H_{10}$ [513-81-5] SDJHPPZKZZWAKF-UHFFFAOYSA-N | $2.0\times10^{-4}$ $2.0\times10^{-4}$ $2.0\times10^{-4}$ $2.0\times10^{-4}$ $2.1\times10^{-4}$ $7.7\times10^{-5}$ $6.0\times10^{-4}$ $1.3\times10^{-4}$ $7.8\times10^{-5}$ $1.9\times10^{-4}$ $8.1\times10^{-6}$ $4.7\times10^{-5}$ $3.8\times10^{-5}$ $9.9\times10^{-5}$ $5.2\times10^{-5}$ $4.7\times10^{-5}$ $8.0\times10^{-5}$ | | Duchowicz et al. (2020) Mackay et al. (2006a) Mackay et al. (1993) Meylan and Howard (1991) Hine and Mookerjee (1975) Yaws (2003) Duchowicz et al. (2020) Gharagheizi et al. (2012) Gharagheizi et al. (2010) Hilal et al. (2008) Modarresi et al. (2005) Yaffe et al. (2003) Yao et al. (2002) Suzuki et al. (1992) Meylan and Howard (1991) Nirmalakhandan and Speece (1988) Yaws (1999) | V V V V V X Q Q Q Q Q Q Q Q Q Q ? | 186 237 246 247 248, 249 229 232 21 |



Table A2.3: Aliphatic alkenes and cycloalkenes (. . . continued)

| Substance Formula (Trivial Name) [CAS Registry Number] InChIKey | $H_s^{cp}$ (at $T^\ominus$) $\left[\dfrac{\mathrm{mol}}{\mathrm{m^3\,Pa}}\right]$ | $\dfrac{\mathrm{d}\ln H_s^{cp}}{\mathrm{d}(1/T)}$ [K] | Reference | Type | Note |
|---|---|---|---|---|---|
| 1,6-heptadiene C$_7$H$_{12}$ [3070-53-9] GEAWFZNTIFJMHR-UHFFFAOYSA-N | $5.6\times10^{-5}$ $4.6\times10^{-5}$ $1.5\times10^{-4}$ | | Plyasunov and Shock (2000) Hilal et al. (2008) Yaffe et al. (2003) | L Q Q | 248, 249 |
| 2,5-dimethyl-2,4-hexadiene C$_8$H$_{14}$ [764-13-6] DZPCYXCBXGQBRN-UHFFFAOYSA-N | $2.1\times10^{-4}$ | | Ebert et al. (2023) | ? | 316 |
| 1-methylcyclopropene C$_4$H$_6$ [3100-04-7] SHDPRTQPPWIEJG-UHFFFAOYSA-N | $2.5\times10^{-4}$ | | HSDB (2015) | Q | 99 |
| cyclopentene C$_5$H$_8$ [142-29-0] | $2.0\times10^{-4}$ | 3300 | Brockbank (2013) | L | 1 |
| | $1.6\times10^{-4}$ | | Plyasunov and Shock (2000) | L | |
| | $2.3\times10^{-4}$ | 2200 | Bakierowska and Trzeszczyński (2003) | M | |
| LPIQUOYDBNQMRZ-UHFFFAOYSA-N | $1.5\times10^{-4}$ | | Duchowicz et al. (2020) | V | 186 |
| | $1.5\times10^{-4}$ | | Mackay et al. (2006a) | V | |
| | $1.5\times10^{-4}$ | | Mackay et al. (1993) | V | |
| | $1.6\times10^{-4}$ | | Hwang et al. (1992) | V | |
| | $1.6\times10^{-4}$ | | Hine and Mookerjee (1975) | V | |
| | $1.5\times10^{-4}$ | | Yaws (2003) | X | 237 |
| | $4.3\times10^{-3}$ | | Duchowicz et al. (2020) | Q | |
| | $1.5\times10^{-4}$ | | Gharagheizi et al. (2012) | Q | |
| | $1.6\times10^{-4}$ | | Raventos-Duran et al. (2010) | Q | 242, 243 |
| | $3.1\times10^{-4}$ | | Raventos-Duran et al. (2010) | Q | 244 |
| | $2.0\times10^{-4}$ | | Raventos-Duran et al. (2010) | Q | 245 |
| | $1.8\times10^{-4}$ | | Gharagheizi et al. (2010) | Q | 246 |
| | $3.1\times10^{-4}$ | | Hilal et al. (2008) | Q | |
| | | 3400 | Kühne et al. (2005) | Q | |
| | $1.5\times10^{-4}$ | | Yaffe et al. (2003) | Q | 248, 249 |
| | $2.8\times10^{-4}$ | | Yao et al. (2002) | Q | 229 |
| | $4.5\times10^{-4}$ | | English and Carroll (2001) | Q | 230, 231 |
| | $1.4\times10^{-4}$ | | Katritzky et al. (1998) | Q | |
| | $1.7\times10^{-4}$ | | Suzuki et al. (1992) | Q | 232 |
| | $1.6\times10^{-4}$ | | Nirmalakhandan and Speece (1988) | Q | |
| | | 2200 | Kühne et al. (2005) | ? | |
| | $1.6\times10^{-4}$ | | Yaws (1999) | ? | 21 |
| | $1.5\times10^{-4}$ | | Yaws and Yang (1992) | ? | 21 |





Table A2.3: Aliphatic alkenes and cycloalkenes (...continued)

| Substance Formula (Trivial Name) [CAS Registry Number] InChIKey | $H_s^{cp}$ (at $T^{\ominus}$) $\left[\dfrac{\text{mol}}{\text{m}^3\,\text{Pa}}\right]$ | $\dfrac{\text{d}\ln H_s^{cp}}{\text{d}(1/T)}$ [K] | Reference | Type | Note |
|---|---|---|---|---|---|
| cyclohexene | $2.9\times10^{-4}$ | 3600 | Brockbank (2013) | L | 1 |
| $C_6H_{10}$ | $2.2\times10^{-4}$ | | Plyasunov and Shock (2000) | L | |
| [110-83-8] | $3.3\times10^{-4}$ | 2000 | Bakierowska and Trzeszczyński (2003) | M | |
| HGCIXCUEYOPUTN-UHFFFAOYSA-N | $2.5\times10^{-4}$ | | Nielsen et al. (1994) | M | |
| | $2.2\times10^{-4}$ | | Mackay et al. (2006a) | V | |
| | $2.2\times10^{-4}$ | | Mackay et al. (1993) | V | |
| | $2.2\times10^{-4}$ | | Hwang et al. (1992) | V | |
| | $2.2\times10^{-4}$ | | Hine and Mookerjee (1975) | V | |
| | $2.2\times10^{-4}$ | | Yaws (2003) | X | 258 |
| | $2.2\times10^{-4}$ | | Yaws (2003) | X | 237 |
| | $3.3\times10^{-4}$ | | Dupeux et al. (2022) | Q | 259 |
| | $6.6\times10^{-5}$ | | Keshavarz et al. (2022) | Q | |
| | $4.3\times10^{-3}$ | | Duchowicz et al. (2020) | Q | 184 |
| | $1.6\times10^{-4}$ | | Gharagheizi et al. (2012) | Q | |
| | $9.9\times10^{-5}$ | | Raventos-Duran et al. (2010) | Q | 242, 243 |
| | $2.5\times10^{-4}$ | | Raventos-Duran et al. (2010) | Q | 244 |
| | $1.6\times10^{-4}$ | | Raventos-Duran et al. (2010) | Q | 245 |
| | $1.5\times10^{-4}$ | | Gharagheizi et al. (2010) | Q | 246 |
| | $2.5\times10^{-4}$ | | Hilal et al. (2008) | Q | |
| | $1.4\times10^{-4}$ | | Modarresi et al. (2007) | Q | 67 |
| | | 3700 | Kühne et al. (2005) | Q | |
| | $2.2\times10^{-4}$ | | Yaffe et al. (2003) | Q | 248, 249 |
| | $2.5\times10^{-4}$ | | Yao et al. (2002) | Q | 229 |
| | $3.3\times10^{-4}$ | | English and Carroll (2001) | Q | 230, 231 |
| | $1.5\times10^{-4}$ | | Katritzky et al. (1998) | Q | |
| | $2.0\times10^{-4}$ | | Russell et al. (1992) | Q | 279 |
| | $1.3\times10^{-4}$ | | Suzuki et al. (1992) | Q | 232 |
| | $1.3\times10^{-4}$ | | Nirmalakhandan and Speece (1988) | Q | |
| | $2.2\times10^{-4}$ | | Duchowicz et al. (2020) | ? | 185, 21 |
| | | 3600 | Kühne et al. (2005) | ? | |
| | $2.2\times10^{-4}$ | | Yaws (1999) | ? | 21 |
| | $2.2\times10^{-4}$ | | Yaws and Yang (1992) | ? | 21 |
| 1-methylcyclopentene | $2.4\times10^{-4}$ | | Hilal et al. (2008) | Q | |
| $C_6H_{10}$ | | | | | |
| [693-89-0] | | | | | |
| ATQUFXWBVZUTKO-UHFFFAOYSA-N | | | | | |
| cycloheptene | $2.0\times10^{-4}$ | | Brockbank (2013) | L | |
| $C_7H_{12}$ | $2.0\times10^{-4}$ | | Plyasunov and Shock (2000) | L | |
| [628-92-2] | $2.6\times10^{-4}$ | | Mackay et al. (2006a) | V | |
| ZXIJMRYMVAMXQP-UHFFFAOYSA-N | $2.0\times10^{-4}$ | | Mackay et al. (1993) | V | |
| | $1.3\times10^{-4}$ | | Hilal et al. (2008) | Q | |



Table A2.3: Aliphatic alkenes and cycloalkenes (. . . continued)

| Substance Formula (Trivial Name) [CAS Registry Number] InChIKey | $H_s^{cp}$ (at $T^\ominus$) $\left[\dfrac{\mathrm{mol}}{\mathrm{m^3\,Pa}}\right]$ | $\dfrac{\mathrm{d}\ln H_s^{cp}}{\mathrm{d}(1/T)}$ [K] | Reference | Type | Note |
|---|---|---|---|---|---|
| 1-methylcyclohexene | $1.4\times10^{-4}$ | | Plyasunov and Shock (2000) | L | |
| $C_6H_9CH_3$ | $1.3\times10^{-4}$ | | Duchowicz et al. (2020) | V | 186 |
| [591-49-1] | $1.2\times10^{-4}$ | | Mackay et al. (2006a) | V | |
| CTMHWPIWNRWQEG-UHFFFAOYSA-N | $1.3\times10^{-4}$ | | Hine and Mookerjee (1975) | V | |
| | $1.4\times10^{-3}$ | | Duchowicz et al. (2020) | Q | |
| | $1.9\times10^{-4}$ | | Hilal et al. (2008) | Q | |
| | $1.2\times10^{-4}$ | | Modarresi et al. (2007) | Q | 67 |
| | $1.3\times10^{-4}$ | | Yaffe et al. (2003) | Q | 248, 249 |
| | $2.4\times10^{-4}$ | | English and Carroll (2001) | Q | 230, 231 |
| | $1.6\times10^{-4}$ | | Katritzky et al. (1998) | Q | |
| | $9.2\times10^{-5}$ | | Suzuki et al. (1992) | Q | 232 |
| cyclooctene | $2.1\times10^{-4}$ | 4300 | Brockbank (2013) | L | 1 |
| $C_8H_{14}$ | $2.1\times10^{-4}$ | 4300 | Doháyosová et al. (2004) | M | 317 |
| [931-88-4] | $2.1\times10^{-4}$ | | Mackay et al. (2006a) | V | |
| URYYVOIYTNXXBN-UPHRSURJSA-N | | 4400 | Kühne et al. (2005) | Q | |
| | | 4400 | Kühne et al. (2005) | ? | |
| 1,1,2,3,3-pentamethyl-2,3,4,5,6,7-hexahydro-1H-indene | $2.5\times10^{-5}$ | | Zhang et al. (2010) | Q | 287, 288 |
| $C_{14}H_{24}$ | $2.5\times10^{-6}$ | | Zhang et al. (2010) | Q | 287, 289 |
| [33704-59-5] | $1.1\times10^{-3}$ | | Zhang et al. (2010) | Q | 287, 290 |
| CDEGOUYLXTUDAU-UHFFFAOYSA-N | $7.3\times10^{-5}$ | | Zhang et al. (2010) | Q | 287, 291 |
| 1,3-cyclopentadiene | $1.8\times10^{-4}$ | | Plyasunov and Shock (2000) | L | |
| $C_5H_6$ | $4.7\times10^{-4}$ | | Duchowicz et al. (2020) | V | 186 |
| [542-92-7] | $4.7\times10^{-4}$ | | HSDB (2015) | V | |
| ZSWFCLXCOIISFI-UHFFFAOYSA-N | $5.1\times10^{-3}$ | | Duchowicz et al. (2020) | Q | |
| | $1.2\times10^{-3}$ | | Hilal et al. (2008) | Q | |
| 1,3-cyclohexadiene | $6.7\times10^{-4}$ | | Yaws (2003) | X | 237 |
| $C_6H_8$ | $8.6\times10^{-4}$ | | Gharagheizi et al. (2012) | Q | |
| [592-57-4] | $4.4\times10^{-4}$ | | Gharagheizi et al. (2010) | Q | 246 |
| MGNZXYYWBUKAII-UHFFFAOYSA-N | $1.1\times10^{-3}$ | | Hilal et al. (2008) | Q | |
| | $6.7\times10^{-4}$ | | Yaws (1999) | ? | 21 |
| 1,4-cyclohexadiene | $1.3\times10^{-3}$ | 4000 | Brockbank (2013) | L | 1 |
| $C_6H_8$ | $1.0\times10^{-3}$ | 3800 | Plyasunov and Shock (2000) | L | |
| (1,4-dihydrobenzene) | $1.1\times10^{-3}$ | | Mackay et al. (2006a) | V | |
| [628-41-1] | $9.7\times10^{-4}$ | | Mackay et al. (1993) | V | |
| UVJHQYIOXKWHFD-UHFFFAOYSA-N | $1.0\times10^{-3}$ | | Hilal et al. (2008) | C | |
| | $4.9\times10^{-4}$ | | Raventos-Duran et al. (2010) | Q | 271, 243 |
| | $6.2\times10^{-4}$ | | Raventos-Duran et al. (2010) | Q | 244 |
| | $9.9\times10^{-5}$ | | Raventos-Duran et al. (2010) | Q | 245 |
| | $8.0\times10^{-4}$ | | Hilal et al. (2008) | Q | |
| | $5.4\times10^{-4}$ | | Modarresi et al. (2007) | Q | 67 |



Table A2.3: Aliphatic alkenes and cycloalkenes (...continued)

| Substance Formula (Trivial Name) [CAS Registry Number] InChIKey | $H_s^{cp}$ (at $T^{\ominus}$) $\left[\dfrac{\mathrm{mol}}{\mathrm{m^3\,Pa}}\right]$ | $\dfrac{\mathrm{d}\ln H_s^{cp}}{\mathrm{d}(1/T)}$ [K] | Reference | Type | Note |
|---|---|---|---|---|---|
| 1,3-cycloheptadiene $C_7H_{10}$ [4054-38-0] GWYPDXLJACEENP-UHFFFAOYSA-N | $6.2\times10^{-4}$ | | Hilal et al. (2008) | Q | |
| 1,3,5-cycloheptatriene $C_7H_8$ [544-25-2] CHVJITGCYZJHLR-UHFFFAOYSA-N | $2.2\times10^{-3}$ $2.1\times10^{-3}$ | 3900 | Plyasunov and Shock (2000) Duchowicz et al. (2020) | L V | 186 |
| | | | Mackay et al. (2006a) | V | 292 |
| | $2.1\times10^{-3}$ | | Mackay et al. (1993) | V | |
| | $2.1\times10^{-3}$ | | Cabani et al. (1981) | V | |
| | $2.2\times10^{-3}$ | | Yaws (2003) | X | 237 |
| | $6.0\times10^{-3}$ | | Duchowicz et al. (2020) | Q | |
| | $4.0\times10^{-3}$ | | Gharagheizi et al. (2012) | Q | |
| | $2.3\times10^{-3}$ | | Gharagheizi et al. (2010) | Q | 246 |
| | $3.8\times10^{-3}$ | | Hilal et al. (2008) | Q | |
| | $3.4\times10^{-4}$ | | Modarresi et al. (2007) | Q | 67 |
| | $3.0\times10^{-3}$ | | English and Carroll (2001) | Q | 230, 231 |
| | $8.4\times10^{-4}$ | | Nirmalakhandan et al. (1997) | Q | |
| | $2.1\times10^{-3}$ | | Yaws (1999) | ? | 21 |
| | $2.1\times10^{-3}$ | | Yaws and Yang (1992) | ? | 21 |
| 1,5-cyclooctadiene $C_8H_{12}$ [111-78-4] VYXHVRARDIDEHS-QGTKBVGQSA-N | $3.8\times10^{-4}$ | | Hilal et al. (2008) | Q | |
| 1-ethenylcyclohexene $C_8H_{12}$ [2622-21-1] SDRZFSPCVYEJTP-UHFFFAOYSA-N | $7.7\times10^{-4}$ | | Hilal et al. (2008) | Q | |
| 4-ethenylcyclohexene $C_8H_{12}$ (4-vinylcyclohexene) [100-40-3] BBDKZWKEPDTENS-UHFFFAOYSA-N | $2.2\times10^{-4}$ $2.2\times10^{-4}$ $2.2\times10^{-4}$ $5.9\times10^{-3}$ $2.5\times10^{-4}$ $1.6\times10^{-4}$ $6.2\times10^{-5}$ $1.8\times10^{-4}$ $2.7\times10^{-4}$ $2.2\times10^{-4}$ $2.7\times10^{-4}$ $2.2\times10^{-4}$ | | Plyasunov and Shock (2000) Duchowicz et al. (2020) HSDB (2015) Duchowicz et al. (2020) Raventos-Duran et al. (2010) Raventos-Duran et al. (2010) Raventos-Duran et al. (2010) Hilal et al. (2008) Modarresi et al. (2007) Yaffe et al. (2003) Katritzky et al. (1998) Yaws (1999) | L V V Q Q Q Q Q Q Q Q ? | 186 271, 243 244 245 67 248, 249 21 |
| 1,3,5,7-cyclooctatetraene $C_8H_8$ [629-20-9] KDUIUFJBNGTBMD-DLMDZQPMSA-N | $3.6\times10^{-2}$ | | Hilal et al. (2008) | Q | |





Table A2.3: Aliphatic alkenes and cycloalkenes (. . . continued)

| Substance Formula (Trivial Name) [CAS Registry Number] InChIKey | $H_s^{cp}$ (at $T^{\ominus}$) $\left[\dfrac{\text{mol}}{\text{m}^3\,\text{Pa}}\right]$ | $\dfrac{\text{d}\ln H_s^{cp}}{\text{d}(1/T)}$ [K] | Reference | Type | Note |
|---|---|---|---|---|---|
| 3a,4,7,7a-tetrahydro-4,7-methano-1H-indene | $1.6\times10^{-4}$ | | HSDB (2015) | Q | 99 |
| $C_{10}H_{12}$ | $2.8\times10^{-5}$ | | Hilal et al. (2008) | Q | |
| (dicyclopentadiene) | | | | | |
| [77-73-6] | | | | | |
| HECLRDQVFMWTQS-UHFFFAOYSA-N | | | | | |
| 1,5,9-cyclododecatriene | $3.3\times10^{-4}$ | | HSDB (2015) | Q | 99 |
| $C_{12}H_{18}$ | | | | | |
| [4904-61-4] | | | | | |
| ZOLLIQAKMYWTBR-UHFFFAOYSA-N | | | | | |
| ($E,E,Z$)-1,5,9-cyclododecatriene | $1.8\times10^{-4}$ | | Ebert et al. (2023) | ? | 318 |
| $C_{12}H_{18}$ | | | | | |
| [706-31-0] | | | | | |
| ZOLLIQAKMYWTBR-RYMQXAEESA-N | | | | | |



### A2.4   Aliphatic alkynes

Table A2.4: Aliphatic alkynes

| Substance Formula (Trivial Name) [CAS Registry Number] InChIKey | $H_s^{cp}$ (at $T^{\ominus}$) $\left[\dfrac{\text{mol}}{\text{m}^3\,\text{Pa}}\right]$ | $\dfrac{\text{d}\ln H_s^{cp}}{\text{d}(1/T)}$ [K] | Reference | Type | Note |
|---|---|---|---|---|---|
| ethyne | $4.1\times10^{-4}$ | 1800 | Burkholder et al. (2019) | L | 1 |
| $C_2H_2$ | $4.1\times10^{-4}$ | 1800 | Burkholder et al. (2015) | L | 1 |
| (acetylene) | $4.1\times10^{-4}$ | 1700 | Sander et al. (2011) | L | 1 |
| [74-86-2] | $4.1\times10^{-4}$ | 1800 | Sander et al. (2006) | L | 1 |
| HSFWRNGVRCDJHI-UHFFFAOYSA-N | $4.1\times10^{-4}$ | 1700 | Fogg et al. (2001) | L | |
| | $4.1\times10^{-4}$ | 1800 | Plyasunov and Shock (2000) | L | |
| | $4.1\times10^{-4}$ | 1800 | Wilhelm et al. (1977) | L | |
| | $4.0\times10^{-4}$ | 2500 | Jadkar and Chaudhari (1980) | M | |
| | $2.5\times10^{-4}$ | | Maillard and Rosenthal (1952) | M | 319 |
| | $3.3\times10^{-4}$ | | Grollman (1929) | M | 58 |
| | $4.2\times10^{-4}$ | 1900 | Gatterer (1926) | M | |
| | $3.7\times10^{-4}$ | | Kremann and Hönel (1913) | M | |
| | $4.1\times10^{-4}$ | 2000 | Winkler (1906) | M | |
| | $5.1\times10^{-4}$ | | Billitzer (1902) | M | 80, 320 |
| | $4.5\times10^{-4}$ | | Duchowicz et al. (2020) | V | 186 |
| | $4.5\times10^{-4}$ | | HSDB (2015) | V | |
| | $3.9\times10^{-4}$ | | Hwang et al. (1992) | V | |
| | $4.1\times10^{-4}$ | | Hine and Mookerjee (1975) | V | |
| | $1.8\times10^{-5}$ | | Pierotti (1965) | T | |
| | $3.9\times10^{-4}$ | | Yaws (2003) | X | 237 |
| | $4.1\times10^{-4}$ | 1800 | Schoen (1923) | X | 321 |
| | $4.4\times10^{-4}$ | | Vítovec (1968) | X | 321, 12 |
| | $4.1\times10^{-4}$ | | Deno and Berkheimer (1960) | C | |
| | $4.2\times10^{-4}$ | | Hayer et al. (2022) | Q | 20 |
| | $8.9\times10^{-3}$ | | Duchowicz et al. (2020) | Q | |
| | $3.5\times10^{-3}$ | | Wang et al. (2017) | Q | 80, 238 |
| | $1.8\times10^{-3}$ | | Wang et al. (2017) | Q | 80, 239 |
| | $2.9\times10^{-3}$ | | Wang et al. (2017) | Q | 80, 240 |
| | $8.4\times10^{-5}$ | | Gharagheizi et al. (2012) | Q | |
| | $3.9\times10^{-4}$ | | Gharagheizi et al. (2010) | Q | 246 |
| | $1.4\times10^{-3}$ | | Hilal et al. (2008) | Q | |
| | | 1800 | Kühne et al. (2005) | Q | |
| | $3.8\times10^{-4}$ | | Suzuki et al. (1992) | Q | 232 |
| | $5.8\times10^{-4}$ | | Nirmalakhandan and Speece (1988) | Q | |
| | $5.0\times10^{-4}$ | | Irmann (1965) | Q | |
| | | 1800 | Kühne et al. (2005) | ? | |
| | $3.9\times10^{-4}$ | | Yaws (1999) | ? | 21 |
| | $4.1\times10^{-4}$ | 1800 | Yaws et al. (1999) | ? | 21 |
| | $4.1\times10^{-4}$ | 1800 | Dean and Lange (1999) | ? | 322, 23 |
| | $3.9\times10^{-4}$ | | Yaws and Yang (1992) | ? | 21 |
| | $4.1\times10^{-4}$ | | Abraham et al. (1990) | ? | |



Table A2.4: Aliphatic alkynes (...continued)

| Substance Formula (Trivial Name) [CAS Registry Number] InChIKey | $H_s^{cp}$ (at $T^\ominus$) $\left[\dfrac{\text{mol}}{\text{m}^3\,\text{Pa}}\right]$ | $\dfrac{\text{d}\ln H_s^{cp}}{\text{d}(1/T)}$ [K] | Reference | Type | Note |
|---|---|---|---|---|---|
| propyne | $7.5\times10^{-4}$ | 2400 | Plyasunov and Shock (2000) | L | |
| CH$_3$CCH | $9.0\times10^{-4}$ | | Mackay and Shiu (1981) | L | |
| [74-99-7] | $9.2\times10^{-4}$ | | McAuliffe (1966) | M | |
| MWWATHDPGQKSAR-UHFFFAOYSA-N | $7.7\times10^{-4}$ | 2500 | Simpson and Lovell (1962) | M | |
| | $6.2\times10^{-4}$ | 2000 | Inga and McKetta (1961) | M | |
| | $9.0\times10^{-4}$ | | Duchowicz et al. (2020) | V | 186 |
| | $9.0\times10^{-4}$ | | HSDB (2015) | V | |
| | $9.0\times10^{-4}$ | | Hine and Mookerjee (1975) | V | |
| | $6.6\times10^{-4}$ | | Irmann (1965) | V | |
| | $9.2\times10^{-4}$ | | Yaws (2003) | X | 237 |
| | $7.6\times10^{-4}$ | | Hayer et al. (2022) | Q | 20 |
| | $3.2\times10^{-3}$ | | Duchowicz et al. (2020) | Q | |
| | $1.1\times10^{-4}$ | | Gharagheizi et al. (2012) | Q | |
| | $9.8\times10^{-4}$ | | Gharagheizi et al. (2010) | Q | 246 |
| | $6.0\times10^{-4}$ | | Hilal et al. (2008) | Q | |
| | $4.0\times10^{-4}$ | | Modarresi et al. (2007) | Q | 67 |
| | | 2100 | Kühne et al. (2005) | Q | |
| | $9.2\times10^{-4}$ | | Yaffe et al. (2003) | Q | 248, 249 |
| | $2.7\times10^{-4}$ | | Suzuki et al. (1992) | Q | 232 |
| | $4.4\times10^{-4}$ | | Nirmalakhandan and Speece (1988) | Q | |
| | $8.4\times10^{-4}$ | | Irmann (1965) | Q | |
| | | 2400 | Kühne et al. (2005) | ? | |
| | $9.3\times10^{-4}$ | | Yaws (1999) | ? | 21 |
| | $9.2\times10^{-4}$ | | Yaws and Yang (1992) | ? | 21 |
| | $9.0\times10^{-4}$ | | Abraham et al. (1990) | ? | |
| | | | Wilhelm et al. (1977) | W | 86 |
| 1-butyne | $5.6\times10^{-3}$ | | Burkholder et al. (2019) | L | |
| C$_2$H$_5$CCH | $5.6\times10^{-4}$ | | Brockbank (2013) | L | |
| (ethylacetylene) | $6.2\times10^{-4}$ | 1900 | Plyasunov and Shock (2000) | L | |
| [107-00-6] | $5.2\times10^{-4}$ | | Mackay and Shiu (1981) | L | |
| KDKYADYSIPSCCQ-UHFFFAOYSA-N | $7.5\times10^{-4}$ | 1900 | Wilhelm et al. (1977) | L | |
| | $5.4\times10^{-4}$ | | McAuliffe (1966) | M | |
| | $7.2\times10^{-4}$ | 1900 | Simpson and Lovell (1962) | M | |
| | $5.2\times10^{-4}$ | | Mackay et al. (2006a) | V | |
| | $2.9\times10^{-4}$ | | Hwang et al. (1992) | V | |
| | $5.3\times10^{-4}$ | | Hine and Mookerjee (1975) | V | |
| | $5.4\times10^{-4}$ | | Yaws (2003) | X | 237 |
| | $8.3\times10^{-5}$ | | Gharagheizi et al. (2012) | Q | |
| | $6.2\times10^{-4}$ | | Gharagheizi et al. (2010) | Q | 246 |
| | $3.6\times10^{-4}$ | | Hilal et al. (2008) | Q | |
| | $2.1\times10^{-4}$ | | Modarresi et al. (2007) | Q | 67 |
| | | 2500 | Kühne et al. (2005) | Q | |
| | $6.9\times10^{-4}$ | | English and Carroll (2001) | Q | 230, 231 |
| | $2.1\times10^{-4}$ | | Suzuki et al. (1992) | Q | 232 |
| | $3.7\times10^{-4}$ | | Nirmalakhandan and Speece (1988) | Q | |
| | $6.4\times10^{-4}$ | | Irmann (1965) | Q | |
| | | 1900 | Kühne et al. (2005) | ? | |



Table A2.4: Aliphatic alkynes (... continued)

| Substance Formula (Trivial Name) [CAS Registry Number] InChIKey | $H_s^{cp}$ (at $T^{\ominus}$) $\left[\dfrac{\text{mol}}{\text{m}^3\,\text{Pa}}\right]$ | $\dfrac{\text{d}\ln H_s^{cp}}{\text{d}(1/T)}$ [K] | Reference | Type | Note |
|---|---|---|---|---|---|
|  | $5.4\times10^{-4}$ |  | Yaws (1999) | ? | 21 |
|  | $5.4\times10^{-4}$ |  | Yaws and Yang (1992) | ? | 21 |
|  | $5.3\times10^{-4}$ |  | Abraham et al. (1990) | ? |  |
| 2-butyne | $4.6\times10^{-4}$ |  | Yaws (2003) | X | 237 |
| $C_4H_6$ | $1.1\times10^{-4}$ |  | Gharagheizi et al. (2012) | Q |  |
| (dimethylacetylene) | $1.3\times10^{-3}$ |  | Gharagheizi et al. (2010) | Q | 246 |
| [503-17-3] | $1.9\times10^{-3}$ |  | Hilal et al. (2008) | Q |  |
| XNMQEEKYCVKGBD-UHFFFAOYSA-N | $4.7\times10^{-4}$ |  | Yaws (1999) | ? | 21 |
| 1-pentyne | $3.9\times10^{-4}$ |  | Brockbank (2013) | L |  |
| $C_3H_7CCH$ | $3.2\times10^{-4}$ |  | Plyasunov and Shock (2000) | L |  |
| [627-19-0] | $4.0\times10^{-4}$ |  | Mackay and Shiu (1981) | L |  |
| IBXNCJKFFQIKKY-UHFFFAOYSA-N | $4.0\times10^{-4}$ |  | Duchowicz et al. (2020) | V | 186 |
|  | $4.0\times10^{-4}$ |  | Mackay et al. (2006a) | V |  |
|  | $4.0\times10^{-4}$ |  | Mackay et al. (1993) | V |  |
|  | $2.5\times10^{-4}$ |  | Amoore and Buttery (1978) | V |  |
|  | $3.9\times10^{-4}$ |  | Hine and Mookerjee (1975) | V |  |
|  | $4.0\times10^{-4}$ |  | Yaws (2003) | X | 237 |
|  | $3.2\times10^{-3}$ |  | Duchowicz et al. (2020) | Q |  |
|  | $6.0\times10^{-5}$ |  | Gharagheizi et al. (2012) | Q |  |
|  | $3.9\times10^{-4}$ |  | Gharagheizi et al. (2010) | Q | 246 |
|  | $2.4\times10^{-4}$ |  | Hilal et al. (2008) | Q |  |
|  | $1.5\times10^{-4}$ |  | Modarresi et al. (2007) | Q | 67 |
|  | $1.3\times10^{-5}$ |  | Modarresi et al. (2005) | Q | 247 |
|  | $4.6\times10^{-4}$ |  | Yaffe et al. (2003) | Q | 248, 249 |
|  | $7.5\times10^{-5}$ |  | Yao et al. (2002) | Q | 229 |
|  | $2.3\times10^{-4}$ |  | English and Carroll (2001) | Q | 230, 260 |
|  | $6.1\times10^{-5}$ |  | Russell et al. (1992) | Q | 279 |
|  | $1.6\times10^{-4}$ |  | Suzuki et al. (1992) | Q | 232 |
|  | $2.9\times10^{-4}$ |  | Nirmalakhandan and Speece (1988) | Q |  |
|  | $4.0\times10^{-4}$ |  | Yaws (1999) | ? | 21 |
|  | $2.0\times10^{-4}$ |  | Yaws and Yang (1992) | ? | 21 |
|  | $3.9\times10^{-4}$ |  | Abraham et al. (1990) | ? |  |
| 2-pentyne | $5.6\times10^{-4}$ |  | Yaws (2003) | X | 237 |
| $C_5H_8$ | $6.7\times10^{-5}$ |  | Gharagheizi et al. (2012) | Q |  |
| [627-21-4] | $7.0\times10^{-4}$ |  | Gharagheizi et al. (2010) | Q | 246 |
| NKTDTMONXHODTI-UHFFFAOYSA-N | $1.1\times10^{-3}$ |  | Hilal et al. (2008) | Q |  |
|  | $1.1\times10^{-5}$ |  | Modarresi et al. (2005) | Q | 247 |
|  | $7.9\times10^{-5}$ |  | Yao et al. (2002) | Q | 229 |
|  | $4.0\times10^{-4}$ |  | Yaws (1999) | ? | 21 |
| 3-methyl-1-butyne | $5.3\times10^{-4}$ |  | Yaws (2003) | X | 237 |
| $C_5H_8$ | $4.4\times10^{-5}$ |  | Gharagheizi et al. (2012) | Q |  |
| [598-23-2] | $6.3\times10^{-4}$ |  | Gharagheizi et al. (2010) | Q | 246 |
| USCSRAJGJYMJFZ-UHFFFAOYSA-N | $3.7\times10^{-4}$ |  | Yaws (1999) | ? | 21 |





Table A2.4: Aliphatic alkynes (...continued)

| Substance Formula (Trivial Name) [CAS Registry Number] InChIKey | $H_s^{cp}$ (at $T^{\ominus}$) $\left[\dfrac{\text{mol}}{\text{m}^3\,\text{Pa}}\right]$ | $\dfrac{\text{d}\ln H_s^{cp}}{\text{d}(1/T)}$ [K] | Reference | Type | Note |
|---|---|---|---|---|---|
| 1-hexyne | $2.6\times10^{-4}$ | | Brockbank (2013) | L | |
| $C_4H_9CCH$ | $3.0\times10^{-4}$ | | Plyasunov and Shock (2000) | L | |
| [693-02-7] | $2.5\times10^{-4}$ | | Duchowicz et al. (2020) | V | 186 |
| CGHIBGNXEGJPQZ-UHFFFAOYSA-N | $2.5\times10^{-4}$ | | Duchowicz et al. (2020) | V | 186 |
| | $2.4\times10^{-4}$ | | Mackay et al. (2006a) | V | |
| | $2.4\times10^{-4}$ | | Mackay et al. (1993) | V | |
| | $2.5\times10^{-4}$ | | Hine and Mookerjee (1975) | V | |
| | $2.5\times10^{-4}$ | | Yaws (2003) | X | 237 |
| | $3.2\times10^{-3}$ | | Duchowicz et al. (2020) | Q | |
| | $3.2\times10^{-3}$ | | Duchowicz et al. (2020) | Q | |
| | $4.8\times10^{-5}$ | | Gharagheizi et al. (2012) | Q | |
| | $2.5\times10^{-4}$ | | Gharagheizi et al. (2010) | Q | 246 |
| | $1.7\times10^{-4}$ | | Hilal et al. (2008) | Q | |
| | $1.5\times10^{-4}$ | | Modarresi et al. (2007) | Q | 67 |
| | $1.1\times10^{-5}$ | | Modarresi et al. (2005) | Q | 247 |
| | $2.4\times10^{-4}$ | | Yaffe et al. (2003) | Q | 248, 249 |
| | $2.7\times10^{-5}$ | | Yao et al. (2002) | Q | 229 |
| | $1.8\times10^{-4}$ | | English and Carroll (2001) | Q | 230, 274 |
| | $1.2\times10^{-4}$ | | Suzuki et al. (1992) | Q | 232 |
| | $2.3\times10^{-4}$ | | Nirmalakhandan and Speece (1988) | Q | |
| | $2.5\times10^{-4}$ | | Yaws (1999) | ? | 21 |
| | $4.6\times10^{-4}$ | | Yaws and Yang (1992) | ? | 21 |
| | $2.5\times10^{-4}$ | | Abraham et al. (1990) | ? | |
| 2-hexyne | $3.5\times10^{-4}$ | | Yaws (2003) | X | 237 |
| $C_6H_{10}$ | $4.5\times10^{-5}$ | | Gharagheizi et al. (2012) | Q | |
| [764-35-2] | $3.8\times10^{-4}$ | | Gharagheizi et al. (2010) | Q | 246 |
| MELUCTCJOARQQG-UHFFFAOYSA-N | $5.8\times10^{-4}$ | | Hilal et al. (2008) | Q | |
| | $1.0\times10^{-5}$ | | Modarresi et al. (2005) | Q | 247 |
| | $3.1\times10^{-5}$ | | Yao et al. (2002) | Q | 229 |
| | $2.5\times10^{-4}$ | | Yaws (1999) | ? | 21 |
| 3-hexyne | $5.5\times10^{-4}$ | | Plyasunov and Shock (2000) | L | |
| $C_6H_{10}$ | $5.6\times10^{-4}$ | | Yaws (2003) | X | 237 |
| [928-49-4] | $4.1\times10^{-5}$ | | Gharagheizi et al. (2012) | Q | |
| DQQNMIPXXNPGCV-UHFFFAOYSA-N | $3.8\times10^{-4}$ | | Gharagheizi et al. (2010) | Q | 246 |
| | $6.0\times10^{-4}$ | | Hilal et al. (2008) | Q | |
| | $1.0\times10^{-5}$ | | Modarresi et al. (2005) | Q | 247 |
| | $3.2\times10^{-5}$ | | Yao et al. (2002) | Q | 229, 267 |
| | $2.6\times10^{-4}$ | | Yaws (1999) | ? | 21 |
| 3-methyl-1-pentyne | $4.6\times10^{-4}$ | | Yaws (2003) | X | 237 |
| $C_6H_{10}$ | $3.5\times10^{-5}$ | | Gharagheizi et al. (2012) | Q | |
| [922-59-8] | $3.9\times10^{-4}$ | | Gharagheizi et al. (2010) | Q | 246 |
| PLHJCCHSCFNKCC-UHFFFAOYSA-N | | | | | |



Table A2.4: Aliphatic alkynes (. . . continued)

| Substance Formula (Trivial Name) [CAS Registry Number] InChIKey | $H_s^{cp}$ (at $T^\ominus$) $\left[\dfrac{\text{mol}}{\text{m}^3\,\text{Pa}}\right]$ | $\dfrac{\text{d}\ln H_s^{cp}}{\text{d}(1/T)}$ [K] | Reference | Type | Note |
|---|---|---|---|---|---|
| 4-methyl-1-pentyne | $4.5\times10^{-4}$ | | Yaws (2003) | X | 237 |
| $C_6H_{10}$ | $3.8\times10^{-5}$ | | Gharagheizi et al. (2012) | Q | |
| [7154-75-8] | $3.9\times10^{-4}$ | | Gharagheizi et al. (2010) | Q | 246 |
| OXRWICUICBZVAE-UHFFFAOYSA-N | | | | | |
| 4-methyl-2-pentyne | $4.1\times10^{-4}$ | | Yaws (2003) | X | 237 |
| $C_6H_{10}$ | $3.8\times10^{-5}$ | | Gharagheizi et al. (2012) | Q | |
| [21020-27-9] | $4.6\times10^{-4}$ | | Gharagheizi et al. (2010) | Q | 246 |
| SLMFWJQZLPEDDU-UHFFFAOYSA-N | | | | | |
| 3,3-dimethyl-1-butyne | $4.6\times10^{-4}$ | | Yaws (2003) | X | 237 |
| $C_6H_{10}$ | $2.0\times10^{-5}$ | | Gharagheizi et al. (2012) | Q | |
| [917-92-0] | $4.8\times10^{-4}$ | | Gharagheizi et al. (2010) | Q | 246 |
| PPWNCLVNXGCGAF-UHFFFAOYSA-N | | | | | |
| 1-heptyne | $1.8\times10^{-4}$ | | Brockbank (2013) | L | |
| $C_5H_{11}CCH$ | $1.8\times10^{-4}$ | | Plyasunov and Shock (2000) | L | |
| [628-71-7] | $1.4\times10^{-4}$ | | Duchowicz et al. (2020) | V | 186 |
| YVXHZKKCZYLQOP-UHFFFAOYSA-N | $1.3\times10^{-4}$ | | Mackay et al. (2006a) | V | |
| | $2.2\times10^{-4}$ | | Mackay et al. (1993) | V | |
| | $1.5\times10^{-4}$ | | Hine and Mookerjee (1975) | V | |
| | $1.4\times10^{-4}$ | | Yaws (2003) | X | 237 |
| | $3.1\times10^{-3}$ | | Duchowicz et al. (2020) | Q | |
| | $4.1\times10^{-5}$ | | Gharagheizi et al. (2012) | Q | |
| | $1.6\times10^{-4}$ | | Gharagheizi et al. (2010) | Q | 246 |
| | $1.1\times10^{-4}$ | | Hilal et al. (2008) | Q | |
| | $1.1\times10^{-4}$ | | Modarresi et al. (2007) | Q | 67 |
| | $9.6\times10^{-6}$ | | Modarresi et al. (2005) | Q | 247 |
| | $1.5\times10^{-4}$ | | Yaffe et al. (2003) | Q | 248, 249 |
| | $2.4\times10^{-5}$ | | Yao et al. (2002) | Q | 229 |
| | $1.3\times10^{-4}$ | | English and Carroll (2001) | Q | 230, 231 |
| | $9.5\times10^{-5}$ | | Suzuki et al. (1992) | Q | 232 |
| | $1.8\times10^{-4}$ | | Nirmalakhandan and Speece (1988) | Q | |
| | $1.4\times10^{-4}$ | | Yaws (1999) | ? | 21 |
| | $1.4\times10^{-4}$ | | Yaws and Yang (1992) | ? | 21 |
| | $1.5\times10^{-4}$ | | Abraham et al. (1990) | ? | |
| 2-heptyne | $4.9\times10^{-4}$ | | Plyasunov and Shock (2000) | L | |
| $C_7H_{12}$ | $5.0\times10^{-4}$ | | Yaws (2003) | X | 237 |
| [1119-65-9] | $4.1\times10^{-5}$ | | Gharagheizi et al. (2012) | Q | |
| AMSFEMSYKQQCHL-UHFFFAOYSA-N | $2.1\times10^{-4}$ | | Gharagheizi et al. (2010) | Q | 246 |
| 3-heptyne | $2.2\times10^{-4}$ | | Yaws (2003) | X | 237 |
| $C_7H_{12}$ | $3.6\times10^{-5}$ | | Gharagheizi et al. (2012) | Q | |
| [2586-89-2] | $2.1\times10^{-4}$ | | Gharagheizi et al. (2010) | Q | 246 |
| KLYHSJRCIZOUHE-UHFFFAOYSA-N | | | | | |





Table A2.4: Aliphatic alkynes (...continued)

| Substance Formula (Trivial Name) [CAS Registry Number] InChIKey | $H_s^{cp}$ (at $T^\ominus$) $\left[\dfrac{\text{mol}}{\text{m}^3\,\text{Pa}}\right]$ | $\dfrac{\text{d}\ln H_s^{cp}}{\text{d}(1/T)}$ [K] | Reference | Type | Note |
|---|---|---|---|---|---|
| 2-methyl-3-hexyne $C_7H_{12}$ [36566-80-0] POBOUPFSQKXZFZ-UHFFFAOYSA-N | $2.2\times10^{-4}$ $2.6\times10^{-4}$ $2.6\times10^{-5}$ $2.5\times10^{-4}$ | | Plyasunov and Shock (2000) Yaws (2003) Gharagheizi et al. (2012) Gharagheizi et al. (2010) | L X Q Q | 237 246 |
| 3-methyl-1-hexyne $C_7H_{12}$ [40276-93-5] OPZULQHRFNTFFZ-UHFFFAOYSA-N | $3.1\times10^{-4}$ $2.7\times10^{-5}$ $2.5\times10^{-4}$ | | Yaws (2003) Gharagheizi et al. (2012) Gharagheizi et al. (2010) | X Q Q | 237 246 |
| 4-methyl-1-hexyne $C_7H_{12}$ [52713-81-2] YFZSGTDENCTWGW-UHFFFAOYSA-N | $2.9\times10^{-4}$ $3.3\times10^{-5}$ $2.5\times10^{-4}$ | | Yaws (2003) Gharagheizi et al. (2012) Gharagheizi et al. (2010) | X Q Q | 237 246 |
| 4-methyl-2-hexyne $C_7H_{12}$ [20198-49-6] ABEXZFRJQJSEBW-UHFFFAOYSA-N | $2.5\times10^{-4}$ $2.5\times10^{-4}$ | | Yaws (2003) Gharagheizi et al. (2010) | X Q | 237 246 |
| 5-methyl-1-hexyne $C_7H_{12}$ [2203-80-7] HKNANEMUCJGPMS-UHFFFAOYSA-N | $2.8\times10^{-4}$ $3.4\times10^{-5}$ $2.5\times10^{-4}$ | | Yaws (2003) Gharagheizi et al. (2012) Gharagheizi et al. (2010) | X Q Q | 237 246 |
| 5-methyl-2-hexyne $C_7H_{12}$ [53566-37-3] SVGAHRUSRQTQES-UHFFFAOYSA-N | $2.4\times10^{-4}$ $3.2\times10^{-5}$ $2.5\times10^{-4}$ | | Yaws (2003) Gharagheizi et al. (2012) Gharagheizi et al. (2010) | X Q Q | 237 246 |
| 3,3-dimethyl-1-pentyne $C_7H_{12}$ [918-82-1] KQIIKSQUHGGYCU-UHFFFAOYSA-N | $3.6\times10^{-4}$ $1.8\times10^{-5}$ $3.0\times10^{-4}$ | | Yaws (2003) Gharagheizi et al. (2012) Gharagheizi et al. (2010) | X Q Q | 237 246 |
| 3,4-dimethyl-1-pentyne $C_7H_{12}$ [61064-08-2] JDQKSTIAVKXRSK-UHFFFAOYSA-N | $3.3\times10^{-4}$ $2.4\times10^{-5}$ $3.8\times10^{-4}$ | | Yaws (2003) Gharagheizi et al. (2012) Gharagheizi et al. (2010) | X Q Q | 237 246 |
| 4,4-dimethyl-1-pentyne $C_7H_{12}$ [13361-63-2] KHBYKPSFBHWBJQ-UHFFFAOYSA-N | $3.3\times10^{-4}$ $2.2\times10^{-5}$ $3.0\times10^{-4}$ | | Yaws (2003) Gharagheizi et al. (2012) Gharagheizi et al. (2010) | X Q Q | 237 246 |
| 4,4-dimethyl-2-pentyne $C_7H_{12}$ [999-78-0] FOALCTWKQSWRST-UHFFFAOYSA-N | $3.0\times10^{-4}$ $1.9\times10^{-5}$ $2.7\times10^{-4}$ | | Yaws (2003) Gharagheizi et al. (2012) Gharagheizi et al. (2010) | X Q Q | 237 246 |



Table A2.4: Aliphatic alkynes (. . . continued)

| Substance Formula (Trivial Name) [CAS Registry Number] InChIKey | $H_s^{cp}$ (at $T^\ominus$) $\left[ \dfrac{\text{mol}}{\text{m}^3\,\text{Pa}} \right]$ | $\dfrac{\text{d}\ln H_s^{cp}}{\text{d}(1/T)}$ [K] | Reference | Type | Note |
|---|---|---|---|---|---|
| 3-ethyl-1-pentyne | $3.2\times10^{-4}$ | | Yaws (2003) | X | 237 |
| $C_7H_{12}$ | $2.7\times10^{-5}$ | | Gharagheizi et al. (2012) | Q | |
| [21020-26-8] | $2.5\times10^{-4}$ | | Gharagheizi et al. (2010) | Q | 246 |
| WGWGXWSBPXLXTA-UHFFFAOYSA-N | | | | | |
| 1-octyne | $1.2\times10^{-4}$ | | Brockbank (2013) | L | |
| $C_6H_{13}CCH$ | $1.2\times10^{-4}$ | | Plyasunov and Shock (2000) | L | |
| [629-05-0] | $1.2\times10^{-4}$ | | Duchowicz et al. (2020) | V | 186 |
| UMIPWJGWASORKV-UHFFFAOYSA-N | $1.3\times10^{-4}$ | | Mackay et al. (2006a) | V | |
| | $1.3\times10^{-4}$ | | Mackay et al. (1993) | V | |
| | $1.2\times10^{-4}$ | | Hine and Mookerjee (1975) | V | |
| | $1.3\times10^{-4}$ | | Yaws (2003) | X | 237 |
| | $3.1\times10^{-3}$ | | Duchowicz et al. (2020) | Q | |
| | $3.1\times10^{-5}$ | | Gharagheizi et al. (2012) | Q | |
| | $1.1\times10^{-4}$ | | Gharagheizi et al. (2010) | Q | 246 |
| | $6.4\times10^{-5}$ | | Hilal et al. (2008) | Q | |
| | $1.0\times10^{-4}$ | | Modarresi et al. (2007) | Q | 67 |
| | $1.2\times10^{-4}$ | | Yaffe et al. (2003) | Q | 248, 249 |
| | $1.0\times10^{-4}$ | | English and Carroll (2001) | Q | 230, 231 |
| | $7.2\times10^{-5}$ | | Suzuki et al. (1992) | Q | 232 |
| | $1.5\times10^{-4}$ | | Nirmalakhandan and Speece (1988) | Q | |
| | $1.2\times10^{-4}$ | | Yaws and Yang (1992) | ? | 21 |
| | $1.2\times10^{-4}$ | | Abraham et al. (1990) | ? | |
| 2-octyne | $1.3\times10^{-4}$ | | Yaws (2003) | X | 237 |
| $C_8H_{14}$ | $3.2\times10^{-5}$ | | Gharagheizi et al. (2012) | Q | |
| [2809-67-8] | $1.3\times10^{-4}$ | | Gharagheizi et al. (2010) | Q | 246 |
| QCQALVMFTWRCFI-UHFFFAOYSA-N | $2.2\times10^{-4}$ | | Hilal et al. (2008) | Q | |
| 3-octyne | $1.5\times10^{-4}$ | | Yaws (2003) | X | 237 |
| $C_8H_{14}$ | $2.8\times10^{-5}$ | | Gharagheizi et al. (2012) | Q | |
| [15232-76-5] | $1.3\times10^{-4}$ | | Gharagheizi et al. (2010) | Q | 246 |
| UDEISTCPVNLKRJ-UHFFFAOYSA-N | | | | | |
| 2,2-dimethyl-3-hexyne | $1.4\times10^{-4}$ | | Plyasunov and Shock (2000) | L | |
| $C_8H_{14}$ | | | | | |
| [4911-60-8] | | | | | |
| XYBFBXTUWDPXLK-UHFFFAOYSA-N | | | | | |
| 1-nonyne | $8.5\times10^{-5}$ | | Brockbank (2013) | L | |
| $C_7H_{15}CCH$ | $9.7\times10^{-5}$ | | Plyasunov and Shock (2000) | L | |
| [3452-09-3] | $6.9\times10^{-5}$ | | Duchowicz et al. (2020) | V | 186 |
| OSSQSXOTMIGBCF-UHFFFAOYSA-N | $6.9\times10^{-5}$ | | Meylan and Howard (1991) | V | |
| | $6.9\times10^{-5}$ | | Hine and Mookerjee (1975) | V | |
| | $7.2\times10^{-5}$ | | Yaws (2003) | X | 237 |
| | $3.1\times10^{-3}$ | | Duchowicz et al. (2020) | Q | |
| | $2.6\times10^{-5}$ | | Gharagheizi et al. (2012) | Q | |
| | $7.8\times10^{-5}$ | | Gharagheizi et al. (2010) | Q | 246 |
| | $4.4\times10^{-5}$ | | Hilal et al. (2008) | Q | |
| | $8.2\times10^{-5}$ | | Modarresi et al. (2007) | Q | 67 |



Table A2.4: Aliphatic alkynes (. . . continued)

| Substance<br>Formula<br>(Trivial Name)<br>[CAS Registry Number]<br>InChIKey | $H_s^{cp}$<br>(at $T^\ominus$)<br>$\left[\dfrac{\mathrm{mol}}{\mathrm{m^3\,Pa}}\right]$ | $\dfrac{\mathrm{d}\ln H_s^{cp}}{\mathrm{d}(1/T)}$<br><br>[K] | Reference | Type | Note |
|---|---|---|---|---|---|
| | $6.9\times10^{-5}$ | | Yaffe et al. (2003) | Q | 248, 249 |
| | $7.9\times10^{-5}$ | | English and Carroll (2001) | Q | 230, 231 |
| | $5.6\times10^{-5}$ | | Suzuki et al. (1992) | Q | 232 |
| | $1.1\times10^{-4}$ | | Meylan and Howard (1991) | Q | |
| | $1.2\times10^{-4}$ | | Nirmalakhandan and Speece (1988) | Q | |
| | $6.9\times10^{-5}$ | | Yaws and Yang (1992) | ? | 21 |
| | $6.9\times10^{-5}$ | | Abraham et al. (1990) | ? | |
| 2-nonyne<br>C$_9$H$_{16}$<br>[19447-29-1]<br>LXKRETAGISZJAD-UHFFFAOYSA-N | $7.3\times10^{-5}$<br>$2.5\times10^{-5}$<br>$8.3\times10^{-5}$ | | Yaws (2003)<br>Gharagheizi et al. (2012)<br>Gharagheizi et al. (2010) | X<br>Q<br>Q | 237<br><br>246 |
| 3-nonyne<br>C$_9$H$_{16}$<br>[20184-89-8]<br>SRRDSRCWRHKEKU-UHFFFAOYSA-N | $7.8\times10^{-5}$<br>$2.2\times10^{-5}$<br>$8.3\times10^{-5}$ | | Yaws (2003)<br>Gharagheizi et al. (2012)<br>Gharagheizi et al. (2010) | X<br>Q<br>Q | 237<br><br>246 |
| 2,2,5-trimethyl-3-hexyne<br>C$_9$H$_{16}$<br>[17530-23-3]<br>DKFPMSNUJIRAMR-UHFFFAOYSA-N | $5.5\times10^{-5}$ | | Plyasunov and Shock (2000) | L | |
| 2,2,5,5-tetramethyl-3-hexyne<br>C$_{10}$H$_{18}$<br>[17530-24-4]<br>FXVDWKZNFZMSOU-UHFFFAOYSA-N | $3.4\times10^{-5}$ | | Plyasunov and Shock (2000) | L | |
| 1-decyne<br>C$_{10}$H$_{18}$<br>[764-93-2]<br>ILLHQJIJCRNRCJ-UHFFFAOYSA-N | $6.6\times10^{-5}$<br>$3.0\times10^{-5}$<br>$6.1\times10^{-5}$ | | Yaws (2003)<br>Gharagheizi et al. (2012)<br>Gharagheizi et al. (2010) | X<br>Q<br>Q | 237<br><br>246 |
| 2-decyne<br>C$_{10}$H$_{18}$<br>[2384-70-5]<br>RWDDSTHSVISBEA-UHFFFAOYSA-N | $5.2\times10^{-5}$<br>$2.9\times10^{-5}$<br>$5.9\times10^{-5}$ | | Yaws (2003)<br>Gharagheizi et al. (2012)<br>Gharagheizi et al. (2010) | X<br>Q<br>Q | 237<br><br>246 |
| 3-decyne<br>C$_{10}$H$_{18}$<br>[2384-85-2]<br>JUWXVJKQNKKRLD-UHFFFAOYSA-N | $5.5\times10^{-5}$<br>$2.5\times10^{-5}$<br>$5.9\times10^{-5}$ | | Yaws (2003)<br>Gharagheizi et al. (2012)<br>Gharagheizi et al. (2010) | X<br>Q<br>Q | 237<br><br>246 |
| 1-undecyne<br>C$_{11}$H$_{20}$<br>[2243-98-3]<br>YVSFLVNWJIEJRV-UHFFFAOYSA-N | $5.9\times10^{-5}$<br>$5.3\times10^{-5}$ | | Yaws (2003)<br>Gharagheizi et al. (2010) | X<br>Q | 237<br>246 |



Table A2.4: Aliphatic alkynes (. . . continued)

| Substance Formula (Trivial Name) [CAS Registry Number] InChIKey | $H_s^{cp}$ (at $T^\ominus$) $\left[\dfrac{\mathrm{mol}}{\mathrm{m^3\,Pa}}\right]$ | $\dfrac{\mathrm{d}\ln H_s^{cp}}{\mathrm{d}(1/T)}$ [K] | Reference | Type | Note |
|---|---|---|---|---|---|
| 2-undecyne $C_{11}H_{20}$ [60212-29-5] XZSXEDPHMIFYOS-UHFFFAOYSA-N | $4.5\times10^{-5}$ $3.3\times10^{-5}$ $4.9\times10^{-5}$ | | Yaws (2003) Gharagheizi et al. (2012) Gharagheizi et al. (2010) | X Q Q | 237 246 |
| 3-undecyne $C_{11}H_{20}$ [60212-30-8] DPWGJNPCPLQVKQ-UHFFFAOYSA-N | $4.6\times10^{-5}$ $2.7\times10^{-5}$ $4.9\times10^{-5}$ | | Yaws (2003) Gharagheizi et al. (2012) Gharagheizi et al. (2010) | X Q Q | 237 246 |
| 1-dodecyne $C_{12}H_{22}$ [765-03-7] ZVDBUOGYYYNMQI-UHFFFAOYSA-N | $5.8\times10^{-5}$ $3.8\times10^{-5}$ $5.4\times10^{-5}$ | | Yaws (2003) Gharagheizi et al. (2012) Gharagheizi et al. (2010) | X Q Q | 237 246 |
| 2-dodecyne $C_{12}H_{22}$ [629-49-2] NDIJGAGRSOPRNJ-UHFFFAOYSA-N | $4.9\times10^{-5}$ $3.6\times10^{-5}$ $4.7\times10^{-5}$ | | Yaws (2003) Gharagheizi et al. (2012) Gharagheizi et al. (2010) | X Q Q | 237 246 |
| 3-dodecyne $C_{12}H_{22}$ [6790-27-8] ZFAGQZXKTQFQLE-UHFFFAOYSA-N | $4.5\times10^{-5}$ $3.0\times10^{-5}$ $4.7\times10^{-5}$ | | Yaws (2003) Gharagheizi et al. (2012) Gharagheizi et al. (2010) | X Q Q | 237 246 |
| 1-tridecyne $C_{13}H_{24}$ [26186-02-7] GZEDKDBFUBPZNG-UHFFFAOYSA-N | $7.4\times10^{-5}$ $4.2\times10^{-5}$ $6.5\times10^{-5}$ | | Yaws (2003) Gharagheizi et al. (2012) Gharagheizi et al. (2010) | X Q Q | 237 246 |
| 2-tridecyne $C_{13}H_{24}$ [28467-75-6] ZGKKGWBQPYIOBH-UHFFFAOYSA-N | $6.7\times10^{-5}$ $3.9\times10^{-5}$ $5.5\times10^{-5}$ | | Yaws (2003) Gharagheizi et al. (2012) Gharagheizi et al. (2010) | X Q Q | 237 246 |
| 3-tridecyne $C_{13}H_{24}$ [60186-78-9] HDLJOUCNTFYRPD-UHFFFAOYSA-N | $5.8\times10^{-5}$ $3.2\times10^{-5}$ $5.5\times10^{-5}$ | | Yaws (2003) Gharagheizi et al. (2012) Gharagheizi et al. (2010) | X Q Q | 237 246 |
| 1-tetradecyne $C_{14}H_{26}$ [765-10-6] DZEFNRWGWQDGTR-UHFFFAOYSA-N | $7.4\times10^{-5}$ $4.4\times10^{-5}$ $9.9\times10^{-5}$ | | Yaws (2003) Gharagheizi et al. (2012) Gharagheizi et al. (2010) | X Q Q | 237 246 |
| 3-tetradecyne $C_{14}H_{26}$ [60212-32-0] DWWMJIDVLGUCDE-UHFFFAOYSA-N | $8.6\times10^{-5}$ $3.3\times10^{-5}$ $8.0\times10^{-5}$ | | Yaws (2003) Gharagheizi et al. (2012) Gharagheizi et al. (2010) | X Q Q | 237 246 |





Table A2.4: Aliphatic alkynes (...continued)

| Substance Formula (Trivial Name) [CAS Registry Number] InChIKey | $H_s^{cp}$ (at $T^{\ominus}$) $\left[\dfrac{\mathrm{mol}}{\mathrm{m^3\,Pa}}\right]$ | $\dfrac{\mathrm{d}\ln H_s^{cp}}{\mathrm{d}(1/T)}$ [K] | Reference | Type | Note |
|---|---|---|---|---|---|
| 1-pentadecyne $C_{15}H_{28}$ [765-13-9] DONJGKADZJEXRJ-UHFFFAOYSA-N | $2.2\times10^{-4}$ $4.5\times10^{-5}$ $1.9\times10^{-4}$ | | Yaws (2003) Gharagheizi et al. (2012) Gharagheizi et al. (2010) | X Q Q | 237 246 |
| 2-pentadecyne $C_{15}H_{28}$ [52112-25-1] VWWAVTXUUYIIEE-UHFFFAOYSA-N | $2.3\times10^{-4}$ $4.4\times10^{-5}$ $1.5\times10^{-4}$ | | Yaws (2003) Gharagheizi et al. (2012) Gharagheizi et al. (2010) | X Q Q | 237 246 |
| 3-pentadecyne $C_{15}H_{28}$ [61886-61-1] RUHQUJVXUGQUNY-UHFFFAOYSA-N | $1.5\times10^{-4}$ $3.4\times10^{-5}$ $1.5\times10^{-4}$ | | Yaws (2003) Gharagheizi et al. (2012) Gharagheizi et al. (2010) | X Q Q | 237 246 |
| 1-hexadecyne $C_{16}H_{30}$ [629-74-3] UCIDYSLOTJMRAM-UHFFFAOYSA-N | $8.3\times10^{-4}$ $5.0\times10^{-4}$ | | Yaws (2003) Gharagheizi et al. (2010) | X Q | 237 246 |
| 3-hexadecyne $C_{16}H_{30}$ [61886-62-2] HRFPRVYNMNOAQO-UHFFFAOYSA-N | $3.3\times10^{-4}$ $3.3\times10^{-5}$ $3.8\times10^{-4}$ | | Yaws (2003) Gharagheizi et al. (2012) Gharagheizi et al. (2010) | X Q Q | 237 246 |
| 3-buten-1-yne $CH_2CHCCH$ (vinylacetylene) [689-97-4] WFYPICNXBKQZGB-UHFFFAOYSA-N | $4.1\times10^{-4}$ $3.7\times10^{-4}$ $3.8\times10^{-4}$ $3.4\times10^{-4}$ $3.4\times10^{-4}$ $1.1\times10^{-2}$ $1.1\times10^{-3}$ $3.5\times10^{-4}$ | 1700 1700 1800 2600 2100 | Plyasunov and Shock (2000) Wilhelm et al. (1977) Simpson and Lovell (1962) Duchowicz et al. (2020) HSDB (2015) Duchowicz et al. (2020) Hilal et al. (2008) Kühne et al. (2005) Kühne et al. (2005) Yaws (1999) | L L M V V Q Q Q ? ? | 323 186 21, 38 |
| butadiyne $C_4H_2$ (biacetylene) [460-12-8] LLCSWKVOHICRDD-UHFFFAOYSA-N | $2.0\times10^{-3}$ $2.7\times10^{-1}$ $8.6\times10^{-3}$ $1.9\times10^{-3}$ $1.9\times10^{-3}$ | | Irmann (1965) Hayer et al. (2022) Hilal et al. (2008) Yaws (1999) Yaws and Yang (1992) | C Q Q ? ? | 20 21 21 |
| 1,6-heptadiyne $C_7H_8$ [2396-63-6] RSPZSDWVQWRAEF-UHFFFAOYSA-N | $4.6\times10^{-3}$ | | Plyasunov and Shock (2000) | L | |
| 1,8-nonadiyne $C_9H_{12}$ [2396-65-8] DMOVPHYFYSASTC-UHFFFAOYSA-N | $3.0\times10^{-3}$ | | Plyasunov and Shock (2000) | L | |





Table A2.4: Aliphatic alkynes (. . . continued)

| Substance<br>Formula<br>(Trivial Name)<br>[CAS Registry Number]<br>InChIKey | $H_s^{cp}$<br>(at $T^\ominus$)<br>$\left[\dfrac{\mathrm{mol}}{\mathrm{m^3\,Pa}}\right]$ | $\dfrac{\mathrm{d}\ln H_s^{cp}}{\mathrm{d}(1/T)}$<br><br>[K] | Reference | Type | Note |
| --- | --- | --- | --- | --- | --- |



### A2.5 Mononuclear aromatics

Table A2.5: Mononuclear aromatics

| Substance Formula (Trivial Name) [CAS Registry Number] InChIKey | $H_s^{cp}$ (at $T^{\ominus}$) $\left[\dfrac{\mathrm{mol}}{\mathrm{m^3\,Pa}}\right]$ | $\dfrac{\mathrm{d\ln} H_s^{cp}}{\mathrm{d}(1/T)}$ [K] | Reference | Type | Note |
|---|---|---|---|---|---|
| benzene | $1.8\times10^{-3}$ | 3800 | Schwardt et al. (2021) | L | 1 |
| $C_6H_6$ | $1.8\times10^{-3}$ | 3700 | Brockbank (2013) | L | 1 |
| [71-43-2] | $1.7\times10^{-3}$ | 4200 | Staudinger and Roberts (2001) | L | |
| UHOVQNZJYSORNB-UHFFFAOYSA-N | $1.8\times10^{-3}$ | 3800 | Plyasunov and Shock (2000) | L | |
| | $1.6\times10^{-3}$ | 4100 | Staudinger and Roberts (1996) | L | |
| | $1.8\times10^{-3}$ | | Mackay and Shiu (1981) | L | |
| | $1.7\times10^{-3}$ | | Kim and Kim (2014) | M | |
| | $1.8\times10^{-3}$ | 3800 | Hiatt (2013) | M | |
| | $2.7\times10^{-3}$ | 1400 | Zhang et al. (2013) | M | 324 |
| | $3.5\times10^{-3}$ | | Zhang et al. (2013) | M | 325 |
| | $1.4\times10^{-3}$ | 2400 | Lau et al. (2010) | M | 11 |
| | $1.7\times10^{-3}$ | 4200 | Sieg et al. (2009) | M | 326 |
| | $1.8\times10^{-3}$ | | Li et al. (2008) | M | |
| | $2.5\times10^{-3}$ | | Lodge and Danso (2007) | M | |
| | $1.4\times10^{-3}$ | 2200 | Lei et al. (2004) | M | 327 |
| | | | Cheng et al. (2003) | M | 328 |
| | $1.8\times10^{-3}$ | | Karl et al. (2003) | M | 87 |
| | $1.8\times10^{-3}$ | 4200 | Bakierowska and Trzeszczyński (2003) | M | |
| | $1.7\times10^{-3}$ | 3800 | Görgényi et al. (2002) | M | 329 |
| | $1.9\times10^{-3}$ | 3200 | Bierwagen and Keller (2001) | M | |
| | $2.1\times10^{-3}$ | | Kochetkov et al. (2001) | M | 297, 330 |
| | $1.7\times10^{-3}$ | | Kochetkov et al. (2001) | M | 297, 331 |
| | $1.8\times10^{-3}$ | | Miller and Stuart (2000) | M | 72 |
| | $3.7\times10^{-3}$ | | Altschuh et al. (1999) | M | |
| | $1.7\times10^{-3}$ | | Ryu and Park (1999) | M | |
| | $1.8\times10^{-3}$ | | Dohnal and Hovorka (1999) | M | |
| | $1.8\times10^{-3}$ | | Allen et al. (1998) | M | |
| | $2.2\times10^{-3}$ | | Peng and Wan (1998) | M | |
| | $1.4\times10^{-3}$ | 3300 | Peng and Wan (1998) | M | 70 |
| | $2.2\times10^{-3}$ | | de Wolf and Lieder (1998) | M | 87 |
| | $1.6\times10^{-3}$ | | Welke et al. (1998) | M | |
| | $1.9\times10^{-3}$ | 3200 | Peng and Wan (1997) | M | |
| | $1.8\times10^{-3}$ | 2700 | Kondoh and Nakajima (1997) | M | |
| | $1.4\times10^{-3}$ | 3300 | Park et al. (1997) | M | |
| | $1.8\times10^{-3}$ | 4200 | Alaee et al. (1996) | M | |
| | $1.6\times10^{-3}$ | 4300 | Turner et al. (1996) | M | |
| | $2.1\times10^{-3}$ | 3900 | Dewulf et al. (1995) | M | |
| | $2.0\times10^{-3}$ | | Nielsen et al. (1994) | M | |
| | $1.7\times10^{-3}$ | 4100 | Khalfaoui and Newsham (1994b) | M | 332 |
| | $1.8\times10^{-3}$ | 3400 | Robbins et al. (1993) | M | 333 |
| | $1.7\times10^{-3}$ | | Hoff et al. (1993) | M | |
| | $1.8\times10^{-3}$ | 2300 | Ettre et al. (1993) | M | 11 |
| | $1.5\times10^{-3}$ | | Hansen et al. (1993) | M | 334 |
| | $1.7\times10^{-3}$ | 4000 | Perlinger et al. (1993) | M | |



Table A2.5: Mononuclear aromatics (...continued)

| Substance<br>Formula<br>(Trivial Name)<br>[CAS Registry Number]<br>InChIKey | $H_s^{cp}$<br>(at $T^{\ominus}$)<br>$\left[\dfrac{\text{mol}}{\text{m}^3\,\text{Pa}}\right]$ | $\dfrac{\text{d}\ln H_s^{cp}}{\text{d}(1/T)}$<br><br>[K] | Reference | Type | Note |
|---|---|---|---|---|---|
| | $1.7\times10^{-3}$ | | Li and Carr (1993) | M | |
| | $1.8\times10^{-3}$ | | Li et al. (1993) | M | |
| | $1.5\times10^{-3}$ | | Zhang and Pawliszyn (1993) | M | |
| | $1.7\times10^{-3}$ | 4300 | Cooling et al. (1992) | M | 335 |
| | $1.8\times10^{-3}$ | | Anderson (1992) | M | 72 |
| | $1.9\times10^{-3}$ | | Yu (1992) | M | 12 |
| | $1.6\times10^{-3}$ | 4300 | Bissonette et al. (1990) | M | |
| | $2.0\times10^{-3}$ | | Guitart et al. (1989) | M | 14 |
| | $1.8\times10^{-3}$ | 3200 | Ashworth et al. (1988) | M | 278 |
| | $1.7\times10^{-3}$ | | Keeley et al. (1988) | M | |
| | $2.0\times10^{-3}$ | | Hellmann (1987) | M | 87 |
| | $1.3\times10^{-3}$ | | Yurteri et al. (1987) | M | 12 |
| | $1.8\times10^{-3}$ | 3600 | Tsonopoulos and Wilson (1983) | M | 1 |
| | $1.7\times10^{-3}$ | 3900 | Sanemasa et al. (1982) | M | |
| | $1.8\times10^{-3}$ | 4000 | Leighton and Calo (1981) | M | |
| | $1.7\times10^{-3}$ | 3500 | Sanemasa et al. (1981) | M | |
| | $1.2\times10^{-3}$ | 5300 | Ervin et al. (1980) | M | |
| | $1.8\times10^{-3}$ | | Warner et al. (1980) | M | |
| | $1.8\times10^{-3}$ | | Mackay et al. (1979) | M | |
| | $1.1\times10^{-3}$ | | Sato and Nakajima (1979a) | M | 14 |
| | $1.6\times10^{-3}$ | 3800 | Tsibul'skii et al. (1979) | M | |
| | $1.8\times10^{-3}$ | 4200 | Green and Frank (1979) | M | |
| | $1.8\times10^{-3}$ | | Vitenberg et al. (1975) | M | |
| | $1.2\times10^{-3}$ | | Vitenberg et al. (1974) | M | 12 |
| | $1.7\times10^{-3}$ | 4400 | Brown and Wasik (1974) | M | |
| | $2.1\times10^{-3}$ | 4500 | Hartkopf and Karger (1973) | M | |
| | $1.6\times10^{-3}$ | 4500 | Wasik and Tsang (1970) | M | |
| | $1.5\times10^{-3}$ | | Saylor et al. (1938) | M | 38 |
| | $3.5\times10^{-4}$ | | Abraham and Acree (2007) | V | |
| | $1.8\times10^{-3}$ | | Mackay et al. (2006a) | V | |
| | $1.8\times10^{-3}$ | | Kochetkov et al. (2001) | V | |
| | $1.8\times10^{-3}$ | | Shiu and Ma (2000) | V | |
| | $1.8\times10^{-3}$ | | Shiu and Mackay (1997) | V | |
| | $1.8\times10^{-3}$ | | Park et al. (1997) | V | |
| | $1.8\times10^{-3}$ | | Mackay et al. (1992a) | V | |
| | $1.8\times10^{-3}$ | | Hwang et al. (1992) | V | |
| | $1.8\times10^{-3}$ | | Eastcott et al. (1988) | V | |
| | $1.8\times10^{-3}$ | 3800 | Abraham (1984) | V | |
| | $1.8\times10^{-3}$ | 3600 | Ben-Naim and Wilf (1980) | V | 1 |
| | $1.8\times10^{-3}$ | | Warner et al. (1980) | V | |
| | $1.8\times10^{-3}$ | | Hine and Mookerjee (1975) | V | |
| | $1.8\times10^{-3}$ | 4100 | Mackay and Leinonen (1975) | V | |
| | $1.7\times10^{-3}$ | 3800 | Wauchope and Haque (1972) | V | |
| | $1.7\times10^{-3}$ | 3800 | Wauchope and Haque (1972) | V | |
| | $2.0\times10^{-3}$ | | McAuliffe (1966) | V | 24 |
| | $1.8\times10^{-3}$ | 3800 | Andon et al. (1954) | V | 336 |
| | $1.8\times10^{-3}$ | | Bohon and Claussen (1951) | V | |



Table A2.5: Mononuclear aromatics (... continued)

| Substance Formula (Trivial Name) [CAS Registry Number] InChIKey | $H_s^{cp}$ (at $T^{\ominus}$) $\left[\dfrac{\text{mol}}{\text{m}^3\,\text{Pa}}\right]$ | $\dfrac{\text{d}\ln H_s^{cp}}{\text{d}(1/T)}$ [K] | Reference | Type | Note |
|---|---|---|---|---|---|
| | $1.8\times10^{-3}$ | 3800 | Plyasunov et al. (2001) | T | |
| | $1.8\times10^{-3}$ | | Mackay et al. (1979) | T | |
| | | 3800 | Gill et al. (1976) | T | |
| | $2.7\times10^{-3}$ | | Pierotti (1965) | T | |
| | $1.8\times10^{-3}$ | | Yaws (2003) | X | 258 |
| | $1.8\times10^{-3}$ | | Yaws (2003) | X | 237 |
| | $1.8\times10^{-3}$ | 2200 | Goldstein (1982) | X | 298 |
| | $1.8\times10^{-3}$ | | Sieg et al. (2008) | C | |
| | $1.8\times10^{-3}$ | | Schüürmann (2000) | C | 21 |
| | $1.8\times10^{-3}$ | | Smith et al. (1993) | C | 12 |
| | $1.8\times10^{-3}$ | | Ryan et al. (1988) | C | |
| | $1.8\times10^{-3}$ | | Shen (1982) | C | |
| | $1.8\times10^{-3}$ | | Dupeux et al. (2022) | Q | 259 |
| | $1.4\times10^{-3}$ | | Hayer et al. (2022) | Q | 20 |
| | $7.2\times10^{-4}$ | | Keshavarz et al. (2022) | Q | |
| | $6.0\times10^{-3}$ | | Duchowicz et al. (2020) | Q | 299 |
| | $6.5\times10^{-3}$ | | Wang et al. (2017) | Q | 80, 238 |
| | $1.3\times10^{-3}$ | | Wang et al. (2017) | Q | 80, 239 |
| | $3.2\times10^{-3}$ | | Wang et al. (2017) | Q | 80, 240 |
| | $1.8\times10^{-3}$ | | Li et al. (2014) | Q | 241 |
| | $4.4\times10^{-3}$ | | Gharagheizi et al. (2012) | Q | |
| | $2.0\times10^{-3}$ | | Raventos-Duran et al. (2010) | Q | 242, 243 |
| | $1.6\times10^{-3}$ | | Raventos-Duran et al. (2010) | Q | 244 |
| | $2.0\times10^{-3}$ | | Raventos-Duran et al. (2010) | Q | 245 |
| | $1.2\times10^{-3}$ | | Gharagheizi et al. (2010) | Q | 246 |
| | $1.7\times10^{-3}$ | | Hilal et al. (2008) | Q | |
| | $2.2\times10^{-3}$ | | Modarresi et al. (2007) | Q | 67 |
| | | 4000 | Kühne et al. (2005) | Q | |
| | $1.8\times10^{-3}$ | | Yaffe et al. (2003) | Q | 248, 249 |
| | $7.4\times10^{-4}$ | | Yao et al. (2002) | Q | 229 |
| | $2.2\times10^{-3}$ | | English and Carroll (2001) | Q | 230, 231 |
| | $7.7\times10^{-5}$ | | Katritzky et al. (1998) | Q | |
| | $2.1\times10^{-3}$ | | Suzuki et al. (1992) | Q | 232 |
| | $2.2\times10^{-3}$ | | Nirmalakhandan and Speece (1988) | Q | |
| | $1.8\times10^{-3}$ | | Arbuckle (1983) | Q | |
| | $1.8\times10^{-3}$ | | Duchowicz et al. (2020) | ? | 185, 21 |
| | | 3700 | Kühne et al. (2005) | ? | |
| | $1.8\times10^{-3}$ | | Yaws (1999) | ? | 21 |
| | $1.1\times10^{-3}$ | | Abraham and Weathersby (1994) | ? | 21 |
| | $1.8\times10^{-3}$ | | Yaws and Yang (1992) | ? | 21 |
| | $1.8\times10^{-3}$ | | Abraham et al. (1990) | ? | |
| | $2.2\times10^{-3}$ | | Mackay and Yeun (1983) | ? | |
| benzene-d6 $C_6D_6$ [1076-43-3] UHOVQNZJYSORNB-MZWXYZOWSA-N | $1.8\times10^{-3}$ $1.6\times10^{-3}$ | 4000 4500 3800 | Hiatt (2013) Wasik and Tsang (1970) Gill et al. (1976) | M M T | |



Table A2.5: Mononuclear aromatics (...continued)

| Substance Formula (Trivial Name) [CAS Registry Number] InChIKey | $H_s^{cp}$ (at $T^\ominus$) $\left[\dfrac{\mathrm{mol}}{\mathrm{m^3\,Pa}}\right]$ | $\dfrac{\mathrm{d}\ln H_s^{cp}}{\mathrm{d}(1/T)}$ [K] | Reference | Type | Note |
|---|---|---|---|---|---|
| methylbenzene | $1.6\times10^{-3}$ | 4100 | Schwardt et al. (2021) | L | 1 |
| $C_6H_5CH_3$ | $1.9\times10^{-3}$ | 4000 | Brockbank (2013) | L | 1, 337 |
| (toluene) | $1.5\times10^{-3}$ | 4300 | Staudinger and Roberts (2001) | L | |
| [108-88-3] | $1.6\times10^{-3}$ | 4400 | Plyasunov and Shock (2000) | L | |
| YXFVVABEGXRONW-UHFFFAOYSA-N | $1.5\times10^{-3}$ | 4000 | Staudinger and Roberts (1996) | L | |
| | $1.5\times10^{-3}$ | | Mackay and Shiu (1981) | L | |
| | $1.5\times10^{-3}$ | 4600 | Kutsuna and Kaneyasu (2021) | M | |
| | $1.5\times10^{-3}$ | | Kim and Kim (2014) | M | |
| | $2.1\times10^{-3}$ | 4400 | Hiatt (2013) | M | |
| | $2.8\times10^{-3}$ | | Zhang et al. (2013) | M | 325 |
| | $1.7\times10^{-3}$ | 4200 | Lee et al. (2013) | M | |
| | $1.5\times10^{-3}$ | | Kish et al. (2013) | M | |
| | $1.3\times10^{-3}$ | 2700 | Lau et al. (2010) | M | 11 |
| | $1.5\times10^{-3}$ | 4300 | Sieg et al. (2009) | M | 326 |
| | $1.4\times10^{-3}$ | | Helburn et al. (2008) | M | |
| | $1.5\times10^{-3}$ | | Li et al. (2008) | M | |
| | $1.3\times10^{-3}$ | 2100 | Falabella and Teja (2008) | M | 11, 338 |
| | $1.4\times10^{-3}$ | | Lodge and Danso (2007) | M | |
| | $1.5\times10^{-3}$ | 3900 | Lin and Chou (2006) | M | |
| | | | Cheng et al. (2004) | M | 328 |
| | $1.4\times10^{-3}$ | 2200 | Lei et al. (2004) | M | 327 |
| | | | Cheng et al. (2003) | M | 328 |
| | $1.4\times10^{-3}$ | | Karl et al. (2003) | M | 87 |
| | $2.1\times10^{-3}$ | | Bobadilla et al. (2003) | M | |
| | $1.7\times10^{-3}$ | 4300 | Bakierowska and Trzeszczyński (2003) | M | |
| | $2.0\times10^{-3}$ | | Destaillats and Charles (2002) | M | |
| | $1.5\times10^{-3}$ | 4200 | Görgényi et al. (2002) | M | 339 |
| | $1.7\times10^{-3}$ | 3600 | Bierwagen and Keller (2001) | M | |
| | $1.0\times10^{-3}$ | | Ayuttaya et al. (2001) | M | 340 |
| | $1.7\times10^{-4}$ | | Ayuttaya et al. (2001) | M | 341 |
| | $7.8\times10^{-4}$ | | Ayuttaya et al. (2001) | M | 342 |
| | $2.3\times10^{-3}$ | | Ayuttaya et al. (2001) | M | 343 |
| | $1.5\times10^{-3}$ | | David et al. (2000) | M | 72 |
| | $1.6\times10^{-3}$ | | Miller and Stuart (2000) | M | 72 |
| | $1.9\times10^{-3}$ | 4000 | Vane and Giroux (2000) | M | |
| | $8.5\times10^{-4}$ | | McIntosh and Heffron (2000) | M | 14 |
| | $1.5\times10^{-3}$ | 4700 | Dewulf et al. (1999) | M | |
| | $1.7\times10^{-3}$ | | Altschuh et al. (1999) | M | |
| | $1.5\times10^{-3}$ | | Ryu and Park (1999) | M | |
| | $1.6\times10^{-3}$ | | Dohnal and Hovorka (1999) | M | |
| | $1.5\times10^{-3}$ | | Allen et al. (1998) | M | |
| | $2.1\times10^{-3}$ | | Peng and Wan (1998) | M | |
| | $1.2\times10^{-3}$ | 3600 | Peng and Wan (1998) | M | 70 |
| | $2.0\times10^{-3}$ | | de Wolf and Lieder (1998) | M | 87 |
| | $1.4\times10^{-3}$ | | Welke et al. (1998) | M | |
| | $1.7\times10^{-3}$ | 3700 | Peng and Wan (1997) | M | |



Table A2.5: Mononuclear aromatics (. . . continued)

| Substance Formula (Trivial Name) [CAS Registry Number] InChIKey | $H_s^{cp}$ (at $T^{\ominus}$) $\left[\dfrac{\text{mol}}{\text{m}^3\,\text{Pa}}\right]$ | $\dfrac{\text{d}\ln H_s^{cp}}{\text{d}(1/T)}$ [K] | Reference | Type | Note |
|---|---|---|---|---|---|
| | $1.7\times10^{-3}$ | 2800 | Kondoh and Nakajima (1997) | M | |
| | $1.3\times10^{-3}$ | 3900 | Park et al. (1997) | M | |
| | $1.4\times10^{-3}$ | 4100 | Turner et al. (1996) | M | |
| | $1.5\times10^{-3}$ | | Ramachandran et al. (1996) | M | |
| | $1.8\times10^{-3}$ | 4400 | Dewulf et al. (1995) | M | |
| | $1.6\times10^{-3}$ | | Nielsen et al. (1994) | M | |
| | $1.5\times10^{-3}$ | 4400 | Robbins et al. (1993) | M | 344 |
| | $1.3\times10^{-3}$ | | Hoff et al. (1993) | M | |
| | $1.5\times10^{-3}$ | 2500 | Ettre et al. (1993) | M | 11 |
| | $1.4\times10^{-3}$ | | Hansen et al. (1993) | M | 334 |
| | $1.5\times10^{-3}$ | 4500 | Perlinger et al. (1993) | M | |
| | $1.6\times10^{-3}$ | | Li and Carr (1993) | M | |
| | $1.6\times10^{-3}$ | | Li et al. (1993) | M | |
| | $1.5\times10^{-3}$ | | Zhang and Pawliszyn (1993) | M | |
| | $1.6\times10^{-3}$ | 2500 | Kolb et al. (1992) | M | 277 |
| | $1.5\times10^{-3}$ | | Anderson (1992) | M | 72 |
| | $3.7\times10^{-3}$ | | Yu (1992) | M | 12 |
| | $1.4\times10^{-3}$ | 5000 | Bissonette et al. (1990) | M | |
| | $1.5\times10^{-3}$ | 6500 | Lamarche and Droste (1989) | M | 345 |
| | $1.5\times10^{-3}$ | 3000 | Ashworth et al. (1988) | M | 278 |
| | $1.6\times10^{-3}$ | | Keeley et al. (1988) | M | |
| | $1.7\times10^{-3}$ | | Yurteri et al. (1987) | M | 12 |
| | $1.2\times10^{-3}$ | 5400 | Schoene and Steinhanses (1985) | M | |
| | $1.5\times10^{-3}$ | | Garbarini and Lion (1985) | M | |
| | $1.5\times10^{-3}$ | 4200 | Sanemasa et al. (1982) | M | |
| | $1.5\times10^{-3}$ | 3800 | Leighton and Calo (1981) | M | |
| | $1.6\times10^{-3}$ | 4100 | Sanemasa et al. (1981) | M | |
| | $1.5\times10^{-3}$ | 4900 | Ervin et al. (1980) | M | |
| | $1.7\times10^{-3}$ | | Warner et al. (1980) | M | |
| | $1.5\times10^{-3}$ | | Mackay et al. (1979) | M | |
| | $8.6\times10^{-4}$ | | Sato and Nakajima (1979a) | M | 14 |
| | $1.5\times10^{-3}$ | 4700 | Tsibul'skii et al. (1979) | M | |
| | $1.9\times10^{-3}$ | | Vitenberg et al. (1975) | M | |
| | $1.6\times10^{-3}$ | 5000 | Brown and Wasik (1974) | M | |
| | $2.0\times10^{-3}$ | 4900 | Hartkopf and Karger (1973) | M | |
| | $1.7\times10^{-3}$ | 5900 | Wasik and Tsang (1970) | M | |
| | $1.6\times10^{-3}$ | | Martins et al. (2017) | V | 315 |
| | $1.5\times10^{-3}$ | | Mackay et al. (2006a) | V | |
| | $1.9\times10^{-3}$ | 4300 | Fogg and Sangster (2003) | V | 346 |
| | $1.5\times10^{-3}$ | | Shiu and Ma (2000) | V | |
| | $1.5\times10^{-3}$ | | Park et al. (1997) | V | |
| | $1.5\times10^{-3}$ | | Mackay et al. (1992a) | V | |
| | $1.3\times10^{-3}$ | | Hwang et al. (1992) | V | |
| | $1.7\times10^{-3}$ | | Eastcott et al. (1988) | V | |
| | $1.5\times10^{-3}$ | 4400 | Abraham (1984) | V | |
| | $1.9\times10^{-3}$ | 4200 | Ben-Naim and Wilf (1980) | V | 1 |
| | $1.5\times10^{-3}$ | | Warner et al. (1980) | V | |





Table A2.5: Mononuclear aromatics (. . . continued)

| Substance Formula (Trivial Name) [CAS Registry Number] InChIKey | $H_s^{cp}$ (at $T^{\ominus}$) $\left[\dfrac{\text{mol}}{\text{m}^3\,\text{Pa}}\right]$ | $\dfrac{\text{d}\ln H_s^{cp}}{\text{d}(1/T)}$ [K] | Reference | Type | Note |
|---|---|---|---|---|---|
| | $1.5\times10^{-3}$ | | Hine and Mookerjee (1975) | V | |
| | $1.5\times10^{-3}$ | | Mackay and Leinonen (1975) | V | |
| | $1.8\times10^{-3}$ | 4400 | Wauchope and Haque (1972) | V | |
| | $1.7\times10^{-3}$ | | McAuliffe (1966) | V | 24 |
| | $1.8\times10^{-3}$ | 4300 | Andon et al. (1954) | V | 336 |
| | $1.8\times10^{-3}$ | | Bohon and Claussen (1951) | V | |
| | $1.6\times10^{-3}$ | 4400 | Plyasunov et al. (2001) | T | |
| | $1.5\times10^{-3}$ | | Mackay et al. (1979) | T | |
| | | 4400 | Gill et al. (1976) | T | |
| | $1.6\times10^{-3}$ | | Yaws (2003) | X | 258 |
| | $1.5\times10^{-3}$ | | Yaws (2003) | X | 237 |
| | $1.5\times10^{-3}$ | 1900 | Goldstein (1982) | X | 298 |
| | $1.5\times10^{-3}$ | | McAuliffe (1971) | X | 347 |
| | $1.5\times10^{-3}$ | | Sieg et al. (2008) | C | |
| | $1.5\times10^{-3}$ | | Schüürmann (2000) | C | 21 |
| | $1.7\times10^{-3}$ | | Smith et al. (1993) | C | 12 |
| | $1.4\times10^{-3}$ | | Ryan et al. (1988) | C | |
| | $1.7\times10^{-3}$ | | Shen (1982) | C | |
| | $1.5\times10^{-3}$ | | Dupeux et al. (2022) | Q | 259 |
| | $1.2\times10^{-3}$ | | Hayer et al. (2022) | Q | 20 |
| | $9.7\times10^{-4}$ | | Keshavarz et al. (2022) | Q | |
| | $3.1\times10^{-3}$ | | Duchowicz et al. (2020) | Q | 299 |
| | $3.8\times10^{-3}$ | | Wang et al. (2017) | Q | 80, 238 |
| | $1.1\times10^{-3}$ | | Wang et al. (2017) | Q | 80, 239 |
| | $3.0\times10^{-3}$ | | Wang et al. (2017) | Q | 80, 240 |
| | $1.5\times10^{-3}$ | | Li et al. (2014) | Q | 241 |
| | $2.0\times10^{-3}$ | | Gharagheizi et al. (2012) | Q | |
| | $1.6\times10^{-3}$ | | Raventos-Duran et al. (2010) | Q | 242, 243 |
| | $1.2\times10^{-3}$ | | Raventos-Duran et al. (2010) | Q | 244 |
| | $1.6\times10^{-3}$ | | Raventos-Duran et al. (2010) | Q | 245 |
| | $1.3\times10^{-3}$ | | Gharagheizi et al. (2010) | Q | 246 |
| | $1.5\times10^{-3}$ | | Hilal et al. (2008) | Q | |
| | $1.2\times10^{-3}$ | | Modarresi et al. (2007) | Q | 67 |
| | | 4300 | Kühne et al. (2005) | Q | |
| | $1.6\times10^{-3}$ | | Yaffe et al. (2003) | Q | 248, 249 |
| | $7.2\times10^{-4}$ | | Yao et al. (2002) | Q | 229 |
| | $1.6\times10^{-3}$ | | English and Carroll (2001) | Q | 230, 231 |
| | $2.7\times10^{-4}$ | | Katritzky et al. (1998) | Q | |
| | $1.5\times10^{-3}$ | | Suzuki et al. (1992) | Q | 232 |
| | $1.6\times10^{-3}$ | | Nirmalakhandan and Speece (1988) | Q | |
| | $1.2\times10^{-3}$ | | Arbuckle (1983) | Q | |
| | $1.5\times10^{-3}$ | | Duchowicz et al. (2020) | ? | 185, 21 |
| | | 4200 | Kühne et al. (2005) | ? | |
| | $1.6\times10^{-3}$ | | Yaws (1999) | ? | 21 |
| | $9.0\times10^{-4}$ | | Abraham and Weathersby (1994) | ? | 21 |
| | $1.5\times10^{-3}$ | | Yaws and Yang (1992) | ? | 21 |
| | $1.5\times10^{-3}$ | | Abraham et al. (1990) | ? | |





Table A2.5: Mononuclear aromatics (...continued)

| Substance / Formula / (Trivial Name) / [CAS Registry Number] / InChIKey | $H_s^{cp}$ (at $T^{\ominus}$) $\left[\dfrac{\text{mol}}{\text{m}^3\,\text{Pa}}\right]$ | $\dfrac{\text{d}\ln H_s^{cp}}{\text{d}(1/T)}$ [K] | Reference | Type | Note |
|---|---|---|---|---|---|
| | $1.9\times10^{-3}$ | | Mackay and Yeun (1983) | ? | |
| methylbenzene-d8 $C_6D_5CD_3$ (toluene-d8) [2037-26-5] YXFVVABEGXRONW-JGUCLWPXSA-N | $2.0\times10^{-3}$ | 4300 | Hiatt (2013) | M | |
| 1,2-dimethylbenzene $C_6H_4(CH_3)_2$ (o-xylene) [95-47-6] CTQNGGLPUBDAKN-UHFFFAOYSA-N | $2.0\times10^{-3}$ | 4600 | Schwardt et al. (2021) | L | 1 |
| | $2.0\times10^{-3}$ | 4700 | Brockbank (2013) | L | 1 |
| | $2.4\times10^{-3}$ | 4200 | Fogg and Sangster (2003) | L | |
| | $2.0\times10^{-3}$ | 4300 | Staudinger and Roberts (2001) | L | |
| | $2.0\times10^{-3}$ | 4400 | Plyasunov and Shock (2000) | L | |
| | $1.9\times10^{-3}$ | 4000 | Staudinger and Roberts (1996) | L | |
| | $2.0\times10^{-3}$ | | Mackay and Shiu (1981) | L | |
| | $1.9\times10^{-3}$ | | Kim and Kim (2014) | M | |
| | $3.2\times10^{-3}$ | 4500 | Hiatt (2013) | M | |
| | $2.7\times10^{-3}$ | 8500 | Zhang et al. (2013) | M | 324 |
| | $2.2\times10^{-3}$ | | Zhang et al. (2013) | M | 325 |
| | $2.0\times10^{-3}$ | 4300 | Sieg et al. (2009) | M | 326 |
| | $2.3\times10^{-3}$ | | Li et al. (2008) | M | |
| | $1.7\times10^{-3}$ | 2500 | Falabella and Teja (2008) | M | 11, 338 |
| | $9.6\times10^{-4}$ | | McIntosh and Heffron (2000) | M | 14 |
| | $2.1\times10^{-3}$ | | Dohnal and Hovorka (1999) | M | |
| | $2.2\times10^{-3}$ | | Welke et al. (1998) | M | |
| | $1.9\times10^{-3}$ | 3400 | Kondoh and Nakajima (1997) | M | |
| | $1.4\times10^{-3}$ | | Turner et al. (1996) | M | |
| | $2.4\times10^{-3}$ | 4500 | Dewulf et al. (1995) | M | |
| | $2.0\times10^{-3}$ | 5800 | Robbins et al. (1993) | M | 348 |
| | $1.9\times10^{-3}$ | | Li and Carr (1993) | M | |
| | $2.1\times10^{-3}$ | | Li et al. (1993) | M | |
| | $2.7\times10^{-3}$ | | Zhang and Pawliszyn (1993) | M | |
| | $1.4\times10^{-3}$ | 3000 | Kolb et al. (1992) | M | 277 |
| | $1.7\times10^{-3}$ | | Anderson (1992) | M | 72 |
| | $2.1\times10^{-3}$ | 5600 | Bissonette et al. (1990) | M | |
| | $1.9\times10^{-3}$ | 3200 | Ashworth et al. (1988) | M | 278 |
| | $2.3\times10^{-3}$ | | Yurteri et al. (1987) | M | 12 |
| | $1.9\times10^{-3}$ | 4500 | Sanemasa et al. (1982) | M | |
| | $1.0\times10^{-3}$ | | Sato and Nakajima (1979a) | M | 14 |
| | $2.9\times10^{-3}$ | 5400 | Wasik and Tsang (1970) | M | |
| | $1.8\times10^{-3}$ | | Mackay et al. (2006a) | V | |
| | $1.8\times10^{-3}$ | | Shiu and Ma (2000) | V | |
| | $1.8\times10^{-3}$ | | Mackay et al. (1992a) | V | |
| | $2.3\times10^{-3}$ | | Eastcott et al. (1988) | V | |
| | $1.8\times10^{-3}$ | | Hine and Mookerjee (1975) | V | |
| | $1.9\times10^{-3}$ | | Mackay and Leinonen (1975) | V | |
| | $1.9\times10^{-3}$ | | McAuliffe (1966) | V | 24 |
| | $2.3\times10^{-3}$ | | Yaws (2003) | X | 237 |





Table A2.5: Mononuclear aromatics (...continued)

| Substance Formula (Trivial Name) [CAS Registry Number] InChIKey | $H_s^{cp}$ (at $T^\ominus$) $\left[\dfrac{\text{mol}}{\text{m}^3\,\text{Pa}}\right]$ | $\dfrac{\text{d}\ln H_s^{cp}}{\text{d}(1/T)}$ [K] | Reference | Type | Note |
|---|---|---|---|---|---|
| | $1.9\times10^{-3}$ | | Sieg et al. (2008) | C | |
| | $1.3\times10^{-3}$ | | Keshavarz et al. (2022) | Q | |
| | $1.6\times10^{-3}$ | | Duchowicz et al. (2020) | Q | |
| | $2.3\times10^{-3}$ | | Wang et al. (2017) | Q | 80, 238 |
| | $1.6\times10^{-3}$ | | Wang et al. (2017) | Q | 80, 239 |
| | $3.2\times10^{-3}$ | | Wang et al. (2017) | Q | 80, 240 |
| | $1.3\times10^{-3}$ | | Gharagheizi et al. (2012) | Q | |
| | $9.9\times10^{-4}$ | | Raventos-Duran et al. (2010) | Q | 242, 243 |
| | $1.6\times10^{-3}$ | | Raventos-Duran et al. (2010) | Q | 244 |
| | $1.6\times10^{-3}$ | | Raventos-Duran et al. (2010) | Q | 245 |
| | $1.9\times10^{-3}$ | | Gharagheizi et al. (2010) | Q | 246 |
| | $2.0\times10^{-3}$ | | Hilal et al. (2008) | Q | |
| | $1.1\times10^{-3}$ | | Modarresi et al. (2007) | Q | 67 |
| | | 4100 | Kühne et al. (2005) | Q | |
| | $1.9\times10^{-3}$ | | Yaffe et al. (2003) | Q | 248, 249 |
| | $1.7\times10^{-3}$ | | English and Carroll (2001) | Q | 230, 274 |
| | $3.8\times10^{-4}$ | | Katritzky et al. (1998) | Q | |
| | $1.0\times10^{-3}$ | | Suzuki et al. (1992) | Q | 232 |
| | $1.1\times10^{-3}$ | | Nirmalakhandan and Speece (1988) | Q | |
| | $1.9\times10^{-3}$ | | Duchowicz et al. (2020) | ? | 185, 21 |
| | | 4100 | Kühne et al. (2005) | ? | |
| | $2.4\times10^{-3}$ | | Yaws (1999) | ? | 21 |
| | $1.1\times10^{-3}$ | | Abraham and Weathersby (1994) | ? | 21 |
| | $2.3\times10^{-3}$ | | Yaws and Yang (1992) | ? | 21 |
| | $1.9\times10^{-3}$ | | Abraham et al. (1990) | ? | |
| 1,2-dimethylbenzene-d10 $C_6D_4(CD_3)_2$ (o-xylene-d10) [56004-61-6] CTQNGGLPUBDAKN-ZGYYUIRESA-N | $3.0\times10^{-3}$ | 4700 | Hiatt (2013) | M | |
| 1,3-dimethylbenzene $C_6H_4(CH_3)_2$ (m-xylene) [108-38-3] IVSZLXZYQVIEFR-UHFFFAOYSA-N | $1.4\times10^{-3}$ | 4100 | Schwardt et al. (2021) | L | 1 |
| | $1.5\times10^{-3}$ | 4300 | Brockbank (2013) | L | 1 |
| | $1.4\times10^{-3}$ | 4200 | Staudinger and Roberts (2001) | L | |
| | $1.4\times10^{-3}$ | 4600 | Plyasunov and Shock (2000) | L | |
| | $1.3\times10^{-3}$ | 4200 | Staudinger and Roberts (1996) | L | |
| | $1.4\times10^{-3}$ | | Mackay and Shiu (1981) | L | |
| | $1.3\times10^{-3}$ | | Kim and Kim (2014) | M | |
| | $1.4\times10^{-3}$ | | Li et al. (2008) | M | |
| | $1.3\times10^{-3}$ | | Karl et al. (2003) | M | 87 |
| | $6.6\times10^{-4}$ | | McIntosh and Heffron (2000) | M | 14 |
| | $1.5\times10^{-3}$ | | Dohnal and Hovorka (1999) | M | |
| | $1.5\times10^{-3}$ | 2900 | Kondoh and Nakajima (1997) | M | |
| | $1.6\times10^{-3}$ | 4300 | Dewulf et al. (1995) | M | |
| | $1.3\times10^{-3}$ | | Li and Carr (1993) | M | |
| | $1.5\times10^{-3}$ | | Li et al. (1993) | M | |
| | $1.4\times10^{-3}$ | 6000 | Bissonette et al. (1990) | M | |



Table A2.5: Mononuclear aromatics (...continued)

| Substance Formula (Trivial Name) [CAS Registry Number] InChIKey | $H_s^{cp}$ (at $T^\ominus$) $\left[\dfrac{\mathrm{mol}}{\mathrm{m^3\,Pa}}\right]$ | $\dfrac{\mathrm{d}\ln H_s^{cp}}{\mathrm{d}(1/T)}$ [K] | Reference | Type | Note |
|---|---|---|---|---|---|
| | $1.3\times10^{-3}$ | 3300 | Ashworth et al. (1988) | M | 278 |
| | $1.4\times10^{-3}$ | 4700 | Sanemasa et al. (1982) | M | |
| | $6.4\times10^{-4}$ | | Sato and Nakajima (1979a) | M | 14 |
| | $1.8\times10^{-3}$ | 4500 | Tsibul'skii et al. (1979) | M | |
| | $1.4\times10^{-3}$ | | Mackay et al. (2006a) | V | |
| | $1.7\times10^{-3}$ | 4300 | Fogg and Sangster (2003) | V | |
| | $1.4\times10^{-3}$ | | Shiu and Ma (2000) | V | |
| | $1.4\times10^{-3}$ | | Mackay et al. (1992a) | V | |
| | $1.4\times10^{-3}$ | | Eastcott et al. (1988) | V | |
| | $1.6\times10^{-3}$ | | Hine and Mookerjee (1975) | V | |
| | $1.6\times10^{-3}$ | 4800 | Wauchope and Haque (1972) | V | |
| | $1.7\times10^{-3}$ | 5000 | Andon et al. (1954) | V | 336 |
| | $1.7\times10^{-3}$ | | Bohon and Claussen (1951) | V | |
| | $1.5\times10^{-3}$ | | Yaws (2003) | X | 237 |
| | $1.4\times10^{-3}$ | | Sieg et al. (2008) | C | |
| | $1.3\times10^{-3}$ | | Keshavarz et al. (2022) | Q | |
| | $1.6\times10^{-3}$ | | Duchowicz et al. (2020) | Q | |
| | $2.3\times10^{-3}$ | | Wang et al. (2017) | Q | 80, 238 |
| | $1.2\times10^{-3}$ | | Wang et al. (2017) | Q | 80, 239 |
| | $2.8\times10^{-3}$ | | Wang et al. (2017) | Q | 80, 240 |
| | $1.0\times10^{-3}$ | | Gharagheizi et al. (2012) | Q | |
| | $9.9\times10^{-4}$ | | Raventos-Duran et al. (2010) | Q | 242, 243 |
| | $1.2\times10^{-3}$ | | Raventos-Duran et al. (2010) | Q | 244 |
| | $1.6\times10^{-3}$ | | Raventos-Duran et al. (2010) | Q | 245 |
| | $1.9\times10^{-3}$ | | Gharagheizi et al. (2010) | Q | 246 |
| | $1.5\times10^{-3}$ | | Hilal et al. (2008) | Q | |
| | $7.5\times10^{-4}$ | | Modarresi et al. (2007) | Q | 67 |
| | | 4700 | Kühne et al. (2005) | Q | |
| | $1.4\times10^{-3}$ | | Yaffe et al. (2003) | Q | 248, 249 |
| | $4.4\times10^{-4}$ | | Yao et al. (2002) | Q | 229 |
| | $1.7\times10^{-3}$ | | English and Carroll (2001) | Q | 230, 231 |
| | $3.8\times10^{-4}$ | | Katritzky et al. (1998) | Q | |
| | $1.4\times10^{-3}$ | | Russell et al. (1992) | Q | 279 |
| | $1.0\times10^{-3}$ | | Suzuki et al. (1992) | Q | 232 |
| | $1.1\times10^{-3}$ | | Nirmalakhandan and Speece (1988) | Q | |
| | $1.4\times10^{-3}$ | | Duchowicz et al. (2020) | ? | 185, 21 |
| | | 4900 | Kühne et al. (2005) | ? | |
| | $1.5\times10^{-3}$ | | Yaws (1999) | ? | 21 |
| | $6.7\times10^{-4}$ | | Abraham and Weathersby (1994) | ? | 21 |
| | $1.5\times10^{-3}$ | | Yaws and Yang (1992) | ? | 21 |
| | $1.3\times10^{-3}$ | | Abraham et al. (1990) | ? | |



Table A2.5: Mononuclear aromatics (. . . continued)

| Substance Formula (Trivial Name) [CAS Registry Number] InChIKey | $H_s^{cp}$ (at $T^{\ominus}$) $\left[\dfrac{\mathrm{mol}}{\mathrm{m}^3\,\mathrm{Pa}}\right]$ | $\dfrac{\mathrm{d}\ln H_s^{cp}}{\mathrm{d}(1/T)}$ [K] | Reference | Type | Note |
|---|---|---|---|---|---|
| 1,4-dimethylbenzene | $1.5\times10^{-3}$ | 5000 | Schwardt et al. (2021) | L | 1 |
| $C_6H_4(CH_3)_2$ | $1.4\times10^{-3}$ | 5000 | Brockbank (2013) | L | 1 |
| (p-xylene) | $1.9\times10^{-3}$ | 4200 | Fogg and Sangster (2003) | L | |
| [106-42-3] | $1.3\times10^{-3}$ | 4000 | Staudinger and Roberts (2001) | L | |
| URLKBWYHVLBVBO-UHFFFAOYSA-N | $1.4\times10^{-3}$ | 4600 | Plyasunov and Shock (2000) | L | |
| | $1.3\times10^{-3}$ | 3800 | Staudinger and Roberts (1996) | L | |
| | $1.4\times10^{-3}$ | | Mackay and Shiu (1981) | L | |
| | $1.0\times10^{-3}$ | 2600 | Schwardt et al. (2021) | M | 349, 11 |
| | $1.3\times10^{-3}$ | | Kim and Kim (2014) | M | |
| | $1.4\times10^{-3}$ | | Li et al. (2008) | M | |
| | $1.6\times10^{-3}$ | 4800 | Lin and Chou (2006) | M | |
| | $2.0\times10^{-3}$ | | Bobadilla et al. (2003) | M | |
| | $6.7\times10^{-4}$ | | McIntosh and Heffron (2000) | M | 14 |
| | $1.4\times10^{-3}$ | | Ryu and Park (1999) | M | |
| | $1.5\times10^{-3}$ | | Dohnal and Hovorka (1999) | M | |
| | $1.5\times10^{-3}$ | 2900 | Kondoh and Nakajima (1997) | M | |
| | $9.8\times10^{-4}$ | 3200 | Park et al. (1997) | M | |
| | $1.7\times10^{-3}$ | 4800 | Dewulf et al. (1995) | M | |
| | $1.2\times10^{-3}$ | 3100 | Hansen et al. (1993) | M | 281 |
| | $1.3\times10^{-3}$ | | Li and Carr (1993) | M | |
| | $1.4\times10^{-3}$ | | Li et al. (1993) | M | |
| | $1.7\times10^{-3}$ | | Zhang and Pawliszyn (1993) | M | |
| | $1.2\times10^{-3}$ | 5300 | Bissonette et al. (1990) | M | |
| | $1.3\times10^{-3}$ | 3500 | Ashworth et al. (1988) | M | 278 |
| | $1.3\times10^{-3}$ | 4800 | Sanemasa et al. (1982) | M | |
| | $6.1\times10^{-4}$ | | Sato and Nakajima (1979a) | M | 14 |
| | $2.3\times10^{-3}$ | 5400 | Wasik and Tsang (1970) | M | |
| | $1.2\times10^{-3}$ | | Martins et al. (2017) | V | 315 |
| | $1.8\times10^{-4}$ | | Abraham and Acree (2007) | V | |
| | $1.7\times10^{-3}$ | | Mackay et al. (2006a) | V | |
| | $1.4\times10^{-3}$ | | Shiu and Ma (2000) | V | |
| | $1.5\times10^{-3}$ | | Park et al. (1997) | V | |
| | $1.7\times10^{-3}$ | | Mackay et al. (1992a) | V | |
| | $1.5\times10^{-3}$ | | Hwang et al. (1992) | V | |
| | $1.8\times10^{-3}$ | | Eastcott et al. (1988) | V | |
| | $1.6\times10^{-3}$ | | Hine and Mookerjee (1975) | V | |
| | $1.6\times10^{-3}$ | 4800 | Wauchope and Haque (1972) | V | |
| | $1.6\times10^{-3}$ | 4900 | Andon et al. (1954) | V | 336 |
| | $1.6\times10^{-3}$ | | Bohon and Claussen (1951) | V | |
| | $1.4\times10^{-3}$ | | Foster et al. (1994) | X | 350 |
| | $1.6\times10^{-3}$ | | Yaws (2003) | X | 237 |
| | $1.4\times10^{-3}$ | | Sieg et al. (2008) | C | |
| | $1.3\times10^{-3}$ | | Schüürmann (2000) | C | 21 |
| | $1.3\times10^{-3}$ | | Keshavarz et al. (2022) | Q | |
| | $1.6\times10^{-3}$ | | Duchowicz et al. (2020) | Q | 184 |
| | $2.3\times10^{-3}$ | | Wang et al. (2017) | Q | 80, 238 |
| | $1.4\times10^{-3}$ | | Wang et al. (2017) | Q | 80, 239 |



Table A2.5: Mononuclear aromatics (. . . continued)

| Substance<br>Formula<br>(Trivial Name)<br>[CAS Registry Number]<br>InChIKey | $H_s^{cp}$<br>(at $T^\ominus$)<br>$\left[\dfrac{\text{mol}}{\text{m}^3\,\text{Pa}}\right]$ | $\dfrac{\text{d}\ln H_s^{cp}}{\text{d}(1/T)}$<br><br>[K] | Reference | Type | Note |
|---|---|---|---|---|---|
| | $2.6\times10^{-3}$ | | Wang et al. (2017) | Q | 80, 240 |
| | $9.8\times10^{-4}$ | | Gharagheizi et al. (2012) | Q | |
| | $9.9\times10^{-4}$ | | Raventos-Duran et al. (2010) | Q | 271, 243 |
| | $1.6\times10^{-3}$ | | Raventos-Duran et al. (2010) | Q | 244 |
| | $1.6\times10^{-3}$ | | Raventos-Duran et al. (2010) | Q | 245 |
| | $1.9\times10^{-3}$ | | Gharagheizi et al. (2010) | Q | 246 |
| | $1.5\times10^{-3}$ | | Hilal et al. (2008) | Q | |
| | $1.1\times10^{-3}$ | | Modarresi et al. (2007) | Q | 67 |
| | | 4700 | Kühne et al. (2005) | Q | |
| | $1.5\times10^{-3}$ | | Yaffe et al. (2003) | Q | 248, 249 |
| | $8.1\times10^{-4}$ | | Yao et al. (2002) | Q | 229 |
| | $1.7\times10^{-3}$ | | English and Carroll (2001) | Q | 230, 231 |
| | $3.9\times10^{-4}$ | | Katritzky et al. (1998) | Q | |
| | $1.0\times10^{-3}$ | | Suzuki et al. (1992) | Q | 232 |
| | $1.1\times10^{-3}$ | | Nirmalakhandan and Speece (1988) | Q | |
| | $1.4\times10^{-3}$ | | Duchowicz et al. (2020) | ? | 185, 21 |
| | | 4500 | Kühne et al. (2005) | ? | |
| | $1.6\times10^{-3}$ | | Yaws (1999) | ? | 21 |
| | $6.4\times10^{-4}$ | | Abraham and Weathersby (1994) | ? | 21 |
| | $1.6\times10^{-3}$ | | Yaws and Yang (1992) | ? | 21 |
| | $1.4\times10^{-3}$ | | Abraham et al. (1990) | ? | |
| 1,2,3-trimethylbenzene<br>$C_6H_3(CH_3)_3$<br>[526-73-8]<br>FYGHSUNMUKGBRK-UHFFFAOYSA-N | $2.4\times10^{-3}$ | 4600 | Brockbank (2013) | L | 1 |
| | $2.7\times10^{-3}$ | 4800 | Fogg and Sangster (2003) | L | |
| | $2.6\times10^{-3}$ | 4500 | Plyasunov and Shock (2000) | L | |
| | $3.1\times10^{-3}$ | | Mackay and Shiu (1981) | L | |
| | $1.1\times10^{-3}$ | | Järnberg and Johanson (1995) | M | 14 |
| | $2.4\times10^{-3}$ | 4500 | Sanemasa et al. (1982) | M | |
| | $2.9\times10^{-3}$ | | Mackay et al. (2006a) | V | |
| | $2.9\times10^{-3}$ | | Shiu and Ma (2000) | V | |
| | $3.1\times10^{-3}$ | | Abraham et al. (1994a) | V | |
| | $2.9\times10^{-3}$ | | Mackay et al. (1992a) | V | |
| | $2.7\times10^{-3}$ | | Eastcott et al. (1988) | V | |
| | $1.3\times10^{-3}$ | | Yaws (2003) | X | 237 |
| | $1.8\times10^{-3}$ | | Keshavarz et al. (2022) | Q | |
| | $7.9\times10^{-4}$ | | Duchowicz et al. (2020) | Q | 299 |
| | $1.4\times10^{-3}$ | | Wang et al. (2017) | Q | 80, 238 |
| | $2.2\times10^{-3}$ | | Wang et al. (2017) | Q | 80, 239 |
| | $3.5\times10^{-3}$ | | Wang et al. (2017) | Q | 80, 240 |
| | $8.8\times10^{-4}$ | | Gharagheizi et al. (2012) | Q | |
| | $7.8\times10^{-4}$ | | Raventos-Duran et al. (2010) | Q | 242, 243 |
| | $2.5\times10^{-3}$ | | Raventos-Duran et al. (2010) | Q | 244 |
| | $1.2\times10^{-3}$ | | Raventos-Duran et al. (2010) | Q | 245 |
| | $1.6\times10^{-3}$ | | Gharagheizi et al. (2010) | Q | 246 |
| | $3.1\times10^{-3}$ | | Hilal et al. (2008) | Q | |
| | $8.2\times10^{-4}$ | | Modarresi et al. (2007) | Q | 67 |
| | | 3900 | Kühne et al. (2005) | Q | |
| | $3.1\times10^{-3}$ | | Yaffe et al. (2003) | Q | 248, 249 |



Table A2.5: Mononuclear aromatics (...continued)

| Substance Formula (Trivial Name) [CAS Registry Number] InChIKey | $H_s^{cp}$ (at $T^{\ominus}$) $\left[\dfrac{\mathrm{mol}}{\mathrm{m^3\,Pa}}\right]$ | $\dfrac{\mathrm{d}\ln H_s^{cp}}{\mathrm{d}(1/T)}$ [K] | Reference | Type | Note |
|---|---|---|---|---|---|
| | $2.0\times10^{-3}$ | | English and Carroll (2001) | Q | 230, 274 |
| | $4.6\times10^{-4}$ | | Katritzky et al. (1998) | Q | |
| | $8.2\times10^{-4}$ | | Nirmalakhandan et al. (1997) | Q | |
| | $2.3\times10^{-3}$ | | Duchowicz et al. (2020) | ? | 185, 21 |
| | | 4400 | Kühne et al. (2005) | ? | |
| | $1.3\times10^{-3}$ | | Yaws (1999) | ? | 21 |
| | $2.7\times10^{-3}$ | | Yaws and Yang (1992) | ? | 21 |
| | $2.1\times10^{-3}$ | | Abraham et al. (1990) | ? | |
| 1,2,4-trimethylbenzene $C_6H_3(CH_3)_3$ [95-63-6] GWHJZXXIDMPWGX-UHFFFAOYSA-N | $1.6\times10^{-3}$ | 4700 | Brockbank (2013) | L | 1 |
| | $1.7\times10^{-3}$ | 3100 | Fogg and Sangster (2003) | L | |
| | $1.6\times10^{-3}$ | 4800 | Plyasunov and Shock (2000) | L | |
| | $1.7\times10^{-3}$ | | Mackay and Shiu (1981) | L | |
| | $3.2\times10^{-3}$ | 5200 | Hiatt (2013) | M | |
| | $1.7\times10^{-3}$ | | Li et al. (2008) | M | |
| | $2.3\times10^{-3}$ | 3600 | Kondoh and Nakajima (1997) | M | |
| | $6.2\times10^{-4}$ | | Järnberg and Johanson (1995) | M | 14 |
| | $1.5\times10^{-3}$ | 4300 | Hansen et al. (1993) | M | 281 |
| | $2.1\times10^{-3}$ | | Yurteri et al. (1987) | M | 12 |
| | $1.6\times10^{-3}$ | 4800 | Sanemasa et al. (1982) | M | |
| | $1.8\times10^{-3}$ | | Mackay et al. (2006a) | V | |
| | $1.8\times10^{-3}$ | | Shiu and Ma (2000) | V | |
| | $1.7\times10^{-3}$ | | Abraham et al. (1994a) | V | |
| | $1.8\times10^{-3}$ | | Mackay et al. (1992a) | V | |
| | $1.6\times10^{-3}$ | | Eastcott et al. (1988) | V | |
| | $1.7\times10^{-3}$ | | Hine and Mookerjee (1975) | V | |
| | $1.4\times10^{-3}$ | | Yaws (2003) | X | 237 |
| | $1.8\times10^{-3}$ | | Keshavarz et al. (2022) | Q | |
| | $7.9\times10^{-4}$ | | Duchowicz et al. (2020) | Q | 299 |
| | $1.4\times10^{-3}$ | | Wang et al. (2017) | Q | 80, 238 |
| | $1.9\times10^{-3}$ | | Wang et al. (2017) | Q | 80, 239 |
| | $2.8\times10^{-3}$ | | Wang et al. (2017) | Q | 80, 240 |
| | $6.3\times10^{-4}$ | | Gharagheizi et al. (2012) | Q | |
| | $7.8\times10^{-4}$ | | Raventos-Duran et al. (2010) | Q | 242, 243 |
| | $2.0\times10^{-3}$ | | Raventos-Duran et al. (2010) | Q | 244 |
| | $1.2\times10^{-3}$ | | Raventos-Duran et al. (2010) | Q | 245 |
| | $1.6\times10^{-3}$ | | Gharagheizi et al. (2010) | Q | 246 |
| | $2.1\times10^{-3}$ | | Hilal et al. (2008) | Q | |
| | $7.5\times10^{-4}$ | | Modarresi et al. (2007) | Q | 67 |
| | | 4500 | Kühne et al. (2005) | Q | |
| | $1.8\times10^{-3}$ | | Yaffe et al. (2003) | Q | 248, 249 |
| | $2.0\times10^{-3}$ | | English and Carroll (2001) | Q | 230, 231 |
| | $4.7\times10^{-4}$ | | Katritzky et al. (1998) | Q | |
| | $7.3\times10^{-4}$ | | Suzuki et al. (1992) | Q | 232 |
| | $8.0\times10^{-4}$ | | Nirmalakhandan and Speece (1988) | Q | |
| | $8.2\times10^{-4}$ | | Arbuckle (1983) | Q | |
| | $1.6\times10^{-3}$ | | Duchowicz et al. (2020) | ? | 185, 21 |
| | | 4700 | Kühne et al. (2005) | ? | |



Table A2.5: Mononuclear aromatics (...continued)

| Substance Formula (Trivial Name) [CAS Registry Number] InChIKey | $H_s^{cp}$ (at $T^{\ominus}$) $\left[\dfrac{\mathrm{mol}}{\mathrm{m^3\,Pa}}\right]$ | $\dfrac{\mathrm{d}\ln H_s^{cp}}{\mathrm{d}(1/T)}$ [K] | Reference | Type | Note |
|---|---|---|---|---|---|
| | $1.5\times10^{-3}$ | | Yaws (1999) | ? | 21 |
| | $1.7\times10^{-3}$ | | Yaws and Yang (1992) | ? | 21 |
| | $1.6\times10^{-3}$ | | Abraham et al. (1990) | ? | |
| 1,3,5-trimethylbenzene | $1.2\times10^{-3}$ | 4400 | Brockbank (2013) | L | 1 |
| $C_6H_3(CH_3)_3$ | $1.3\times10^{-3}$ | 4900 | Plyasunov and Shock (2000) | L | |
| (mesitylene) | $1.7\times10^{-3}$ | | Mackay and Shiu (1981) | L | |
| [108-67-8] | $2.3\times10^{-3}$ | 5100 | Hiatt (2013) | M | |
| AUHZEENZYGFFBQ-UHFFFAOYSA-N | $2.0\times10^{-3}$ | | Karl et al. (2003) | M | 87 |
| | $1.5\times10^{-3}$ | 3000 | Kondoh and Nakajima (1997) | M | |
| | $4.8\times10^{-4}$ | | Järnberg and Johanson (1995) | M | 14 |
| | $1.3\times10^{-3}$ | | Li and Carr (1993) | M | |
| | $1.4\times10^{-3}$ | | Li et al. (1993) | M | |
| | $1.4\times10^{-3}$ | 3600 | Ashworth et al. (1988) | M | 278 |
| | $1.1\times10^{-3}$ | 4700 | Sanemasa et al. (1982) | M | |
| | $1.1\times10^{-3}$ | 4600 | Sanemasa et al. (1981) | M | |
| | $1.4\times10^{-4}$ | | Abraham and Acree (2007) | V | |
| | $1.3\times10^{-3}$ | | Mackay et al. (2006a) | V | |
| | $1.3\times10^{-3}$ | | Shiu and Ma (2000) | V | |
| | $1.8\times10^{-3}$ | | Abraham et al. (1994a) | V | |
| | $1.3\times10^{-3}$ | | Mackay et al. (1992a) | V | |
| | $1.2\times10^{-3}$ | | Eastcott et al. (1988) | V | |
| | $1.2\times10^{-3}$ | | Yaws (2003) | X | 237 |
| | $9.1\times10^{-4}$ | | Hayer et al. (2022) | Q | 20 |
| | $1.8\times10^{-3}$ | | Keshavarz et al. (2022) | Q | |
| | $7.9\times10^{-4}$ | | Duchowicz et al. (2020) | Q | 299 |
| | $1.4\times10^{-3}$ | | Wang et al. (2017) | Q | 80, 238 |
| | $1.1\times10^{-3}$ | | Wang et al. (2017) | Q | 80, 239 |
| | $2.5\times10^{-3}$ | | Wang et al. (2017) | Q | 80, 240 |
| | $5.2\times10^{-4}$ | | Gharagheizi et al. (2012) | Q | |
| | $1.6\times10^{-3}$ | | Gharagheizi et al. (2010) | Q | 246 |
| | $1.4\times10^{-3}$ | | Hilal et al. (2008) | Q | |
| | $9.0\times10^{-4}$ | | Modarresi et al. (2007) | Q | 67 |
| | | 5000 | Kühne et al. (2005) | Q | |
| | $1.8\times10^{-3}$ | | Yaffe et al. (2003) | Q | 248, 249 |
| | $4.6\times10^{-4}$ | | Katritzky et al. (1998) | Q | |
| | $8.0\times10^{-4}$ | | Nirmalakhandan et al. (1997) | Q | |
| | $1.1\times10^{-3}$ | | Duchowicz et al. (2020) | ? | 185, 21 |
| | | 4400 | Kühne et al. (2005) | ? | |
| | $1.2\times10^{-3}$ | | Yaws (1999) | ? | 21 |
| | $1.2\times10^{-3}$ | | Yaws and Yang (1992) | ? | 21 |
| | $1.3\times10^{-3}$ | | Abraham et al. (1990) | ? | |
| 1,2,3,4-tetramethylbenzene $C_{10}H_{14}$ [488-23-3] UOHMMEJUHBCKEE-UHFFFAOYSA-N | $2.9\times10^{-3}$ | 5900 | Brockbank (2013) | L | 1 |



Table A2.5: Mononuclear aromatics (...continued)

| Substance Formula (Trivial Name) [CAS Registry Number] InChIKey | $H_s^{cp}$ (at $T^{\ominus}$) $\left[\dfrac{\text{mol}}{\text{m}^3\,\text{Pa}}\right]$ | $\dfrac{\text{d}\ln H_s^{cp}}{\text{d}(1/T)}$ [K] | Reference | Type | Note |
|---|---|---|---|---|---|
| 1,2,3,5-tetramethylbenzene $C_{10}H_{14}$ [527-53-7] BFIMMTCNYPIMRN-UHFFFAOYSA-N | $1.2\times10^{-3}$ | | Zhang et al. (2010) | Q | 287, 288 |
| | $2.2\times10^{-3}$ | | Zhang et al. (2010) | Q | 287, 289 |
| | $2.2\times10^{-3}$ | | Zhang et al. (2010) | Q | 287, 290 |
| | $4.1\times10^{-4}$ | | Zhang et al. (2010) | Q | 287, 291 |
| | $1.2\times10^{-3}$ | | Yaws (1999) | ? | 21 |
| 1,2,4,5-tetramethylbenzene $C_{10}H_{14}$ [95-93-2] SQNZJJAZBFDUTD-UHFFFAOYSA-N | $1.4\times10^{-3}$ | | Brockbank (2013) | L | |
| | $1.3\times10^{-3}$ | | Plyasunov and Shock (2000) | L | |
| | $3.9\times10^{-4}$ | | Mackay and Shiu (1981) | L | |
| | $3.9\times10^{-4}$ | | Mackay et al. (2006a) | V | |
| | $3.9\times10^{-4}$ | | Mackay et al. (1992a) | V | |
| | $3.9\times10^{-4}$ | | Eastcott et al. (1988) | V | |
| | $3.5\times10^{-4}$ | | Yaws (2003) | X | 237 |
| | $1.4\times10^{-3}$ | | Abraham et al. (2019) | Q | |
| | $3.7\times10^{-4}$ | | Gharagheizi et al. (2012) | Q | |
| | $1.2\times10^{-3}$ | | Zhang et al. (2010) | Q | 287, 288 |
| | $2.5\times10^{-3}$ | | Zhang et al. (2010) | Q | 287, 289 |
| | $1.9\times10^{-3}$ | | Zhang et al. (2010) | Q | 287, 290 |
| | $4.1\times10^{-4}$ | | Zhang et al. (2010) | Q | 287, 291 |
| | $3.7\times10^{-4}$ | | Gharagheizi et al. (2010) | Q | 246 |
| | $2.9\times10^{-3}$ | | Hilal et al. (2008) | Q | |
| | $8.2\times10^{-4}$ | | Modarresi et al. (2007) | Q | 67 |
| | $3.9\times10^{-4}$ | | Yaffe et al. (2003) | Q | 248, 249 |
| | $3.9\times10^{-4}$ | | Yaws and Yang (1992) | ? | 21 |
| ethylbenzene $C_6H_5C_2H_5$ [100-41-4] YNQLUTRBYVCPMQ-UHFFFAOYSA-N | $1.4\times10^{-3}$ | 4500 | Schwardt et al. (2021) | L | 1, 351 |
| | $1.3\times10^{-3}$ | 5000 | Brockbank (2013) | L | 1 |
| | $1.4\times10^{-3}$ | 4800 | Fogg and Sangster (2003) | L | |
| | $1.3\times10^{-3}$ | 5100 | Staudinger and Roberts (2001) | L | |
| | $1.3\times10^{-3}$ | 4800 | Plyasunov and Shock (2000) | L | |
| | $1.2\times10^{-3}$ | 5100 | Staudinger and Roberts (1996) | L | |
| | $1.3\times10^{-3}$ | | Mackay and Shiu (1981) | L | |
| | $1.3\times10^{-3}$ | 4400 | Schwardt et al. (2021) | M | 352 |
| | $2.0\times10^{-3}$ | 4100 | Hiatt (2013) | M | |
| | $1.9\times10^{-3}$ | 4200 | Zhang et al. (2013) | M | 324 |
| | $1.4\times10^{-3}$ | | Zhang et al. (2013) | M | 325 |
| | $1.3\times10^{-3}$ | 5100 | Sieg et al. (2009) | M | 326 |
| | $1.4\times10^{-3}$ | | Li et al. (2008) | M | |
| | $1.2\times10^{-3}$ | 2700 | Falabella and Teja (2008) | M | 11, 338 |
| | $1.1\times10^{-3}$ | | Lodge and Danso (2007) | M | |
| | | | Cheng et al. (2003) | M | 328 |
| | $1.6\times10^{-3}$ | | Miller and Stuart (2000) | M | 72 |
| | $1.1\times10^{-3}$ | | Ryu and Park (1999) | M | 353 |
| | $1.3\times10^{-3}$ | | Allen et al. (1998) | M | |
| | $1.4\times10^{-3}$ | 2800 | Kondoh and Nakajima (1997) | M | |
| | $1.1\times10^{-3}$ | | Turner et al. (1996) | M | |
| | $1.5\times10^{-3}$ | 4900 | Dewulf et al. (1995) | M | |
| | $1.3\times10^{-3}$ | 5000 | Robbins et al. (1993) | M | 354 |





Table A2.5: Mononuclear aromatics (...continued)

| Substance Formula (Trivial Name) [CAS Registry Number] InChIKey | $H_s^{cp}$ (at $T^{\ominus}$) $\left[\dfrac{\mathrm{mol}}{\mathrm{m^3\,Pa}}\right]$ | $\dfrac{\mathrm{d}\ln H_s^{cp}}{\mathrm{d}(1/T)}$ [K] | Reference | Type | Note |
|---|---|---|---|---|---|
| | $1.3\times10^{-3}$ | 5300 | Perlinger et al. (1993) | M | |
| | $1.3\times10^{-3}$ | | Li and Carr (1993) | M | |
| | $1.3\times10^{-3}$ | | Li et al. (1993) | M | |
| | $2.5\times10^{-3}$ | | Zhang and Pawliszyn (1993) | M | |
| | $1.1\times10^{-3}$ | 5500 | Bissonette et al. (1990) | M | |
| | $1.2\times10^{-3}$ | 5000 | Ashworth et al. (1988) | M | 278 |
| | $1.3\times10^{-3}$ | 4400 | Heidman et al. (1985) | M | 1 |
| | $1.3\times10^{-3}$ | 4600 | Sanemasa et al. (1982) | M | |
| | $1.4\times10^{-3}$ | 4500 | Sanemasa et al. (1981) | M | |
| | $1.4\times10^{-3}$ | 5500 | Ervin et al. (1980) | M | |
| | $1.5\times10^{-3}$ | | Warner et al. (1980) | M | |
| | $1.2\times10^{-3}$ | | Mackay et al. (1979) | M | |
| | $6.6\times10^{-4}$ | | Sato and Nakajima (1979a) | M | 14 |
| | $1.3\times10^{-3}$ | 5600 | Brown and Wasik (1974) | M | |
| | $1.6\times10^{-3}$ | 6400 | Hartkopf and Karger (1973) | M | |
| | $1.6\times10^{-4}$ | | Abraham and Acree (2007) | V | |
| | $1.1\times10^{-3}$ | | Mackay et al. (2006a) | V | |
| | $1.2\times10^{-3}$ | | Shiu and Ma (2000) | V | |
| | $1.2\times10^{-3}$ | | Lide and Frederikse (1995) | V | |
| | $1.1\times10^{-3}$ | | Mackay et al. (1992a) | V | |
| | $1.2\times10^{-3}$ | | Hwang et al. (1992) | V | |
| | $1.0\times10^{-3}$ | | Eastcott et al. (1988) | V | |
| | $1.2\times10^{-3}$ | 4800 | Abraham (1984) | V | |
| | $1.6\times10^{-3}$ | 4900 | Ben-Naim and Wilf (1980) | V | 1 |
| | $1.5\times10^{-3}$ | | Warner et al. (1980) | V | |
| | $1.1\times10^{-3}$ | | Hine and Mookerjee (1975) | V | |
| | $1.5\times10^{-3}$ | 4800 | Wauchope and Haque (1972) | V | |
| | $1.3\times10^{-3}$ | | McAuliffe (1966) | V | 24 |
| | $1.5\times10^{-3}$ | 4900 | Andon et al. (1954) | V | 336 |
| | $1.5\times10^{-3}$ | | Bohon and Claussen (1951) | V | |
| | $1.4\times10^{-3}$ | 4900 | Owens et al. (1986) | T | |
| | $1.1\times10^{-3}$ | | Mackay et al. (1979) | T | |
| | | 4800 | Gill et al. (1976) | T | |
| | $1.2\times10^{-3}$ | | Yaws (2003) | X | 237 |
| | $1.6\times10^{-3}$ | 1700 | Goldstein (1982) | X | 298 |
| | $1.3\times10^{-3}$ | | Sieg et al. (2008) | C | |
| | $1.6\times10^{-3}$ | | Ryan et al. (1988) | C | |
| | $1.5\times10^{-3}$ | | Shen (1982) | C | |
| | $9.7\times10^{-4}$ | | Hayer et al. (2022) | Q | 20 |
| | $1.3\times10^{-3}$ | | Keshavarz et al. (2022) | Q | |
| | $3.1\times10^{-3}$ | | Duchowicz et al. (2020) | Q | |
| | $3.1\times10^{-3}$ | | Wang et al. (2017) | Q | 80, 238 |
| | $9.3\times10^{-4}$ | | Wang et al. (2017) | Q | 80, 239 |
| | $2.8\times10^{-3}$ | | Wang et al. (2017) | Q | 80, 240 |
| | $1.4\times10^{-3}$ | | Gharagheizi et al. (2012) | Q | |
| | $9.9\times10^{-4}$ | | Raventos-Duran et al. (2010) | Q | 242, 243 |
| | $9.9\times10^{-4}$ | | Raventos-Duran et al. (2010) | Q | 244 |



Table A2.5: Mononuclear aromatics (. . . continued)

| Substance Formula (Trivial Name) [CAS Registry Number] InChIKey | $H_s^{cp}$ (at $T^\ominus$) $\left[\dfrac{\mathrm{mol}}{\mathrm{m}^3\,\mathrm{Pa}}\right]$ | $\dfrac{\mathrm{d}\ln H_s^{cp}}{\mathrm{d}(1/T)}$ [K] | Reference | Type | Note |
|---|---|---|---|---|---|
| | $1.2\times10^{-3}$ | | Raventos-Duran et al. (2010) | Q | 245 |
| | $1.1\times10^{-3}$ | | Gharagheizi et al. (2010) | Q | 246 |
| | $1.4\times10^{-3}$ | | Hilal et al. (2008) | Q | |
| | $9.6\times10^{-4}$ | | Modarresi et al. (2007) | Q | 67 |
| | | 4700 | Kühne et al. (2005) | Q | |
| | $1.2\times10^{-3}$ | | Yaffe et al. (2003) | Q | 248, 249 |
| | $5.8\times10^{-4}$ | | Yao et al. (2002) | Q | 229 |
| | $1.4\times10^{-3}$ | | English and Carroll (2001) | Q | 230, 260 |
| | $4.1\times10^{-4}$ | | Katritzky et al. (1998) | Q | |
| | $1.6\times10^{-3}$ | | Russell et al. (1992) | Q | 279 |
| | $1.1\times10^{-3}$ | | Suzuki et al. (1992) | Q | 232 |
| | $1.3\times10^{-3}$ | | Nirmalakhandan and Speece (1988) | Q | |
| | $1.3\times10^{-3}$ | | Arbuckle (1983) | Q | |
| | $1.3\times10^{-3}$ | | Duchowicz et al. (2020) | ? | 185, 21 |
| | | 5000 | Kühne et al. (2005) | ? | |
| | $1.2\times10^{-3}$ | | Yaws (1999) | ? | 21 |
| | $6.9\times10^{-4}$ | | Abraham and Weatherby (1994) | ? | 21 |
| | $1.1\times10^{-3}$ | | Hoff et al. (1993) | ? | 21 |
| | $1.2\times10^{-3}$ | | Yaws and Yang (1992) | ? | 21 |
| | $1.2\times10^{-3}$ | | Abraham et al. (1990) | ? | |
| ethylbenzene-d10 $C_6D_5C_2D_5$ [25837-05-2] YNQLUTRBYVCPMQ-CFTAVCBPSA-N | $2.0\times10^{-3}$ | 4200 | Hiatt (2013) | M | |
| 1,2-diethylbenzene $C_{10}H_{14}$ (*o*-diethylbenzene) [135-01-3] KVNYFPKFSJIPBJ-UHFFFAOYSA-N | $3.8\times10^{-3}$ | | Duchowicz et al. (2020) | V | 186 |
| | $3.8\times10^{-3}$ | | HSDB (2015) | V | |
| | $1.3\times10^{-3}$ | | Yaws (2003) | X | 237 |
| | $1.2\times10^{-3}$ | | Hilal et al. (2008) | C | |
| | $1.6\times10^{-3}$ | | Duchowicz et al. (2020) | Q | |
| | $5.2\times10^{-4}$ | | Gharagheizi et al. (2012) | Q | |
| | $6.2\times10^{-4}$ | | Raventos-Duran et al. (2010) | Q | 271, 243 |
| | $9.9\times10^{-4}$ | | Raventos-Duran et al. (2010) | Q | 244 |
| | $7.8\times10^{-4}$ | | Raventos-Duran et al. (2010) | Q | 245 |
| | $1.1\times10^{-3}$ | | Gharagheizi et al. (2010) | Q | 246 |
| | $1.3\times10^{-3}$ | | Hilal et al. (2008) | Q | |
| | $1.2\times10^{-3}$ | | Modarresi et al. (2007) | Q | 67 |
| | | 4800 | Kühne et al. (2005) | Q | |
| | $3.8\times10^{-3}$ | | Yaffe et al. (2003) | Q | 248, 249 |
| | $5.6\times10^{-4}$ | | Katritzky et al. (1998) | Q | |
| | | 5100 | Kühne et al. (2005) | ? | |
| | $1.3\times10^{-3}$ | | Yaws (1999) | ? | 21 |



Table A2.5: Mononuclear aromatics (. . . continued)

| Substance Formula (Trivial Name) [CAS Registry Number] InChIKey | $H_s^{cp}$ (at $T^\ominus$) $\left[\dfrac{\mathrm{mol}}{\mathrm{m^3\,Pa}}\right]$ | $\dfrac{\mathrm{d}\ln H_s^{cp}}{\mathrm{d}(1/T)}$ [K] | Reference | Type | Note |
|---|---|---|---|---|---|
| 1,3-diethylbenzene | $1.2\times10^{-3}$ | | Duchowicz et al. (2020) | V | 186 |
| $C_{10}H_{14}$ | $1.1\times10^{-3}$ | | HSDB (2015) | V | |
| ($m$-diethylbenzene) | $1.3\times10^{-3}$ | | Yaws (2003) | X | 237 |
| [141-93-5] | $3.8\times10^{-3}$ | | Hilal et al. (2008) | C | |
| AFZZYIJIWUTJFO-UHFFFAOYSA-N | $1.6\times10^{-3}$ | | Duchowicz et al. (2020) | Q | |
| | $4.8\times10^{-4}$ | | Gharagheizi et al. (2012) | Q | |
| | $1.1\times10^{-3}$ | | Gharagheizi et al. (2010) | Q | 246 |
| | $9.7\times10^{-4}$ | | Hilal et al. (2008) | Q | |
| | $4.7\times10^{-4}$ | | Modarresi et al. (2007) | Q | 67 |
| | | 5300 | Kühne et al. (2005) | Q | |
| | $5.6\times10^{-4}$ | | Katritzky et al. (1998) | Q | |
| | | 5300 | Kühne et al. (2005) | ? | |
| | $1.3\times10^{-3}$ | | Yaws (1999) | ? | 21 |
| 1,4-diethylbenzene | $1.3\times10^{-3}$ | | Duchowicz et al. (2020) | V | 186 |
| $C_{10}H_{14}$ | $1.4\times10^{-3}$ | | HSDB (2015) | V | |
| ($p$-diethylbenzene) | $1.2\times10^{-3}$ | | Yaws (2003) | X | 237 |
| [105-05-5] | $1.6\times10^{-3}$ | | Duchowicz et al. (2020) | Q | |
| DSNHSQKRULAAEI-UHFFFAOYSA-N | $5.0\times10^{-4}$ | | Gharagheizi et al. (2012) | Q | |
| | $1.1\times10^{-3}$ | | Gharagheizi et al. (2010) | Q | 246 |
| | $1.1\times10^{-3}$ | | Hilal et al. (2008) | Q | |
| | $7.6\times10^{-4}$ | | Modarresi et al. (2007) | Q | 67 |
| | | 5300 | Kühne et al. (2005) | Q | |
| | $1.4\times10^{-3}$ | | Yaffe et al. (2003) | Q | 248, 249 |
| | $5.7\times10^{-4}$ | | Katritzky et al. (1998) | Q | |
| | $7.9\times10^{-4}$ | | Nirmalakhandan et al. (1997) | Q | |
| | | 5900 | Kühne et al. (2005) | ? | |
| | $1.2\times10^{-3}$ | | Yaws (1999) | ? | 21 |
| | | | Brockbank (2013) | W | 355 |
| propylbenzene | $9.0\times10^{-4}$ | 3800 | Schwardt et al. (2021) | L | 1 |
| $C_6H_5C_3H_7$ | $9.1\times10^{-4}$ | 4700 | Brockbank (2013) | L | 1 |
| [103-65-1] | $9.1\times10^{-4}$ | 5300 | Plyasunov and Shock (2000) | L | |
| ODLMAHJVESYWTB-UHFFFAOYSA-N | $1.4\times10^{-3}$ | | Mackay and Shiu (1981) | L | |
| | $1.9\times10^{-3}$ | 4500 | Hiatt (2013) | M | |
| | $1.5\times10^{-3}$ | | Karl et al. (2003) | M | 87 |
| | $1.1\times10^{-3}$ | 2600 | Kondoh and Nakajima (1997) | M | |
| | $8.6\times10^{-4}$ | 5400 | Perlinger et al. (1993) | M | |
| | $9.3\times10^{-4}$ | | Li and Carr (1993) | M | |
| | $9.1\times10^{-4}$ | | Li et al. (1993) | M | |
| | $9.0\times10^{-4}$ | 3700 | Ashworth et al. (1988) | M | 278 |
| | $9.5\times10^{-4}$ | 4700 | Sanemasa et al. (1982) | M | |
| | $5.0\times10^{-4}$ | | Sato and Nakajima (1979a) | M | 14 |
| | $9.6\times10^{-4}$ | | Mackay et al. (2006a) | V | |
| | $9.6\times10^{-4}$ | | Shiu and Ma (2000) | V | |
| | $9.6\times10^{-4}$ | | Mackay et al. (1992a) | V | |
| | $9.7\times10^{-4}$ | | Eastcott et al. (1988) | V | |
| | $9.7\times10^{-4}$ | 5300 | Abraham (1984) | V | |





Table A2.5: Mononuclear aromatics (. . . continued)

| Substance Formula (Trivial Name) [CAS Registry Number] InChIKey | $H_s^{cp}$ (at $T^{\ominus}$) $\left[\dfrac{\text{mol}}{\text{m}^3\,\text{Pa}}\right]$ | $\dfrac{\text{d}\ln H_s^{cp}}{\text{d}(1/T)}$ [K] | Reference | Type | Note |
|---|---|---|---|---|---|
| | $1.5\times10^{-3}$ | 5500 | Ben-Naim and Wilf (1980) | V | 1 |
| | $9.9\times10^{-4}$ | | Hine and Mookerjee (1975) | V | |
| | $9.6\times10^{-4}$ | 5100 | Owens et al. (1986) | T | |
| | | 5300 | Gill et al. (1976) | T | |
| | $7.4\times10^{-4}$ | | Yaws (2003) | X | 237 |
| | $1.8\times10^{-3}$ | | Keshavarz et al. (2022) | Q | |
| | $3.1\times10^{-3}$ | | Duchowicz et al. (2020) | Q | |
| | $2.5\times10^{-3}$ | | Wang et al. (2017) | Q | 80, 238 |
| | $6.6\times10^{-4}$ | | Wang et al. (2017) | Q | 80, 239 |
| | $2.6\times10^{-3}$ | | Wang et al. (2017) | Q | 80, 240 |
| | $9.9\times10^{-4}$ | | Li et al. (2014) | Q | 241 |
| | $1.0\times10^{-3}$ | | Gharagheizi et al. (2012) | Q | |
| | $7.8\times10^{-4}$ | | Raventos-Duran et al. (2010) | Q | 242, 243 |
| | $7.8\times10^{-4}$ | | Raventos-Duran et al. (2010) | Q | 244 |
| | $9.9\times10^{-4}$ | | Raventos-Duran et al. (2010) | Q | 245 |
| | $8.6\times10^{-4}$ | | Gharagheizi et al. (2010) | Q | 246 |
| | $9.9\times10^{-4}$ | | Hilal et al. (2008) | Q | |
| | $8.6\times10^{-4}$ | | Modarresi et al. (2007) | Q | 67 |
| | | 5000 | Kühne et al. (2005) | Q | |
| | $1.0\times10^{-3}$ | | Yaffe et al. (2003) | Q | 248, 272 |
| | $1.1\times10^{-3}$ | | English and Carroll (2001) | Q | 230, 231 |
| | $5.1\times10^{-4}$ | | Katritzky et al. (1998) | Q | |
| | $1.3\times10^{-3}$ | | Russell et al. (1992) | Q | 279 |
| | $8.8\times10^{-4}$ | | Suzuki et al. (1992) | Q | 232 |
| | $1.1\times10^{-3}$ | | Nirmalakhandan and Speece (1988) | Q | |
| | $9.4\times10^{-4}$ | | Duchowicz et al. (2020) | ? | 185, 21 |
| | | 4700 | Kühne et al. (2005) | ? | |
| | $1.7\times10^{-3}$ | | Yaws (1999) | ? | 21 |
| | $5.2\times10^{-4}$ | | Abraham and Weathersby (1994) | ? | 21 |
| | $9.6\times10^{-4}$ | | Yaws and Yang (1992) | ? | 21 |
| | $9.7\times10^{-4}$ | | Abraham et al. (1990) | ? | |
| (2-propyl)-benzene $C_6H_5C_3H_7$ (isopropylbenzene; cumene) [98-82-8] RWGFKTVRMDUZSP-UHFFFAOYSA-N | $9.0\times10^{-4}$ | 4800 | Brockbank (2013) | L | 1 |
| | $1.2\times10^{-3}$ | 3200 | Staudinger and Roberts (2001) | L | |
| | $8.4\times10^{-4}$ | 4800 | Plyasunov and Shock (2000) | L | |
| | $7.7\times10^{-3}$ | | Mackay and Shiu (1981) | L | |
| | $1.4\times10^{-3}$ | 4900 | Hiatt (2013) | M | |
| | $1.0\times10^{-3}$ | 2500 | Kondoh and Nakajima (1997) | M | |
| | $8.7\times10^{-4}$ | 3300 | Hansen et al. (1993) | M | 281 |
| | $9.1\times10^{-4}$ | | Li and Carr (1993) | M | |
| | $8.9\times10^{-4}$ | | Li et al. (1993) | M | |
| | $1.6\times10^{-3}$ | 3200 | Ashworth et al. (1988) | M | 278 |
| | $8.9\times10^{-4}$ | 4700 | Sanemasa et al. (1982) | M | |
| | $5.6\times10^{-4}$ | | Sato and Nakajima (1979a) | M | 14 |
| | $6.8\times10^{-4}$ | | Mackay et al. (2006a) | V | |
| | $6.8\times10^{-4}$ | | Shiu and Ma (2000) | V | |
| | $6.8\times10^{-4}$ | | Mackay et al. (1992a) | V | |
| | $6.8\times10^{-4}$ | | Hwang et al. (1992) | V | |



Table A2.5: Mononuclear aromatics (...continued)

| Substance Formula (Trivial Name) [CAS Registry Number] InChIKey | $H_s^{cp}$ (at $T^{\ominus}$) $\left[\dfrac{\mathrm{mol}}{\mathrm{m}^3\,\mathrm{Pa}}\right]$ | $\dfrac{\mathrm{d}\ln H_s^{cp}}{\mathrm{d}(1/T)}$ [K] | Reference | Type | Note |
|---|---|---|---|---|---|
| | $6.6\times10^{-4}$ | | Eastcott et al. (1988) | V | |
| | $6.7\times10^{-4}$ | | Hine and Mookerjee (1975) | V | |
| | $6.8\times10^{-4}$ | | Mackay and Leinonen (1975) | V | |
| | $1.1\times10^{-3}$ | 5000 | Wauchope and Haque (1972) | V | |
| | $1.1\times10^{-3}$ | | McAuliffe (1966) | V | 24 |
| | $6.8\times10^{-4}$ | | Yaws (2003) | X | 237 |
| | $1.8\times10^{-3}$ | | Keshavarz et al. (2022) | Q | |
| | $1.2\times10^{-3}$ | | Duchowicz et al. (2020) | Q | 299 |
| | $2.9\times10^{-3}$ | | Wang et al. (2017) | Q | 80, 238 |
| | $5.3\times10^{-4}$ | | Wang et al. (2017) | Q | 80, 239 |
| | $2.8\times10^{-3}$ | | Wang et al. (2017) | Q | 80, 240 |
| | $9.4\times10^{-4}$ | | Savary et al. (2014) | Q | |
| | $8.2\times10^{-4}$ | | Gharagheizi et al. (2012) | Q | |
| | $7.8\times10^{-4}$ | | Raventos-Duran et al. (2010) | Q | 242, 243 |
| | $6.2\times10^{-4}$ | | Raventos-Duran et al. (2010) | Q | 244 |
| | $9.9\times10^{-4}$ | | Raventos-Duran et al. (2010) | Q | 245 |
| | $8.8\times10^{-4}$ | | Gharagheizi et al. (2010) | Q | 246 |
| | $8.6\times10^{-4}$ | | Hilal et al. (2008) | Q | |
| | $7.7\times10^{-4}$ | | Modarresi et al. (2007) | Q | 67 |
| | | 5000 | Kühne et al. (2005) | Q | |
| | | | Yaffe et al. (2003) | Q | 356 |
| | $3.6\times10^{-4}$ | | Yao et al. (2002) | Q | 229 |
| | $1.2\times10^{-3}$ | | English and Carroll (2001) | Q | 230, 231 |
| | $4.7\times10^{-4}$ | | Katritzky et al. (1998) | Q | |
| | $9.2\times10^{-4}$ | | Nirmalakhandan et al. (1997) | Q | |
| | $8.0\times10^{-4}$ | | Suzuki et al. (1992) | Q | 232 |
| | $9.2\times10^{-4}$ | | Nirmalakhandan and Speece (1988) | Q | |
| | $8.6\times10^{-4}$ | | Duchowicz et al. (2020) | ? | 185, 21 |
| | | 4400 | Kühne et al. (2005) | ? | |
| | $6.9\times10^{-4}$ | | Yaws (1999) | ? | 21 |
| | $5.8\times10^{-4}$ | | Abraham and Weathersby (1994) | ? | 21 |
| | $6.8\times10^{-4}$ | | Hoff et al. (1993) | ? | 21 |
| | $6.8\times10^{-4}$ | | Yaws and Yang (1992) | ? | 21 |
| | $8.8\times10^{-4}$ | | Abraham et al. (1990) | ? | |
| | | | Fogg and Sangster (2003) | W | 357 |
| 1-ethyl-2-methylbenzene $C_6H_4CH_3C_2H_5$ (o-ethyltoluene) [611-14-3] HYFLWBNQFMXCPA-UHFFFAOYSA-N | $1.8\times10^{-3}$ | | Plyasunov and Shock (2000) | L | |
| | $2.3\times10^{-3}$ | | Mackay and Shiu (1981) | L | |
| | $1.8\times10^{-3}$ | | Duchowicz et al. (2020) | V | 186 |
| | $1.9\times10^{-3}$ | | Mackay et al. (2006a) | V | |
| | $1.9\times10^{-3}$ | | Mackay et al. (1992a) | V | |
| | $1.9\times10^{-3}$ | | Eastcott et al. (1988) | V | |
| | $2.2\times10^{-3}$ | | Yaws (2003) | X | 237 |
| | $1.6\times10^{-3}$ | | Duchowicz et al. (2020) | Q | |
| | $1.8\times10^{-3}$ | | Wang et al. (2017) | Q | 80, 238 |
| | $1.2\times10^{-3}$ | | Wang et al. (2017) | Q | 80, 239 |
| | $3.0\times10^{-3}$ | | Wang et al. (2017) | Q | 80, 240 |
| | $7.8\times10^{-4}$ | | Raventos-Duran et al. (2010) | Q | 271, 243 |



Table A2.5: Mononuclear aromatics (. . . continued)

| Substance<br>Formula<br>(Trivial Name)<br>[CAS Registry Number]<br>InChIKey | $H_s^{cp}$<br>(at $T^\ominus$)<br>$\left[\dfrac{\mathrm{mol}}{\mathrm{m}^3\,\mathrm{Pa}}\right]$ | $\dfrac{\mathrm{d}\ln H_s^{cp}}{\mathrm{d}(1/T)}$<br><br>[K] | Reference | Type | Note |
|---|---|---|---|---|---|
| | $1.6\times10^{-3}$ | | Raventos-Duran et al. (2010) | Q | 244 |
| | $1.2\times10^{-3}$ | | Raventos-Duran et al. (2010) | Q | 245 |
| | $1.5\times10^{-3}$ | | Gharagheizi et al. (2010) | Q | 246 |
| | $1.8\times10^{-3}$ | | Hilal et al. (2008) | Q | |
| | $8.9\times10^{-4}$ | | Modarresi et al. (2007) | Q | 67 |
| | | 4500 | Kühne et al. (2005) | Q | |
| | $2.4\times10^{-3}$ | | Yaffe et al. (2003) | Q | 248, 249 |
| | $5.1\times10^{-4}$ | | Katritzky et al. (1998) | Q | |
| | $9.5\times10^{-4}$ | | Nirmalakhandan et al. (1997) | Q | |
| | | 3200 | Kühne et al. (2005) | ? | |
| | $2.2\times10^{-3}$ | | Yaws (1999) | ? | 21 |
| | $2.3\times10^{-3}$ | | Yaws and Yang (1992) | ? | 21 |
| 1-ethyl-3-methylbenzene<br>$C_6H_4CH_3C_2H_5$<br>($m$-ethyltoluene)<br>[620-14-4]<br>ZLCSFXXPPANWQY-UHFFFAOYSA-N | $1.7\times10^{-3}$ | | Yaws (2003) | X | 237 |
| | $1.8\times10^{-3}$ | | Wang et al. (2017) | Q | 80, 238 |
| | $9.8\times10^{-4}$ | | Wang et al. (2017) | Q | 80, 239 |
| | $2.6\times10^{-3}$ | | Wang et al. (2017) | Q | 80, 240 |
| | $1.5\times10^{-3}$ | | Gharagheizi et al. (2010) | Q | 246 |
| | $1.3\times10^{-3}$ | | Hilal et al. (2008) | Q | |
| | $1.7\times10^{-3}$ | | Yaws (1999) | ? | 21 |
| 1-ethyl-4-methylbenzene<br>$C_6H_4CH_3C_2H_5$<br>($p$-ethyltoluene)<br>[622-96-8]<br>JRLPEMVDPFPYPJ-UHFFFAOYSA-N | $2.0\times10^{-3}$ | | Mackay and Shiu (1981) | L | |
| | $2.0\times10^{-3}$ | | Duchowicz et al. (2020) | V | 186 |
| | $1.6\times10^{-3}$ | | Duchowicz et al. (2020) | V | 186 |
| | $2.0\times10^{-3}$ | | Mackay et al. (2006a) | V | |
| | $2.0\times10^{-3}$ | | Mackay et al. (1992a) | V | |
| | $2.0\times10^{-3}$ | | Eastcott et al. (1988) | V | |
| | $2.0\times10^{-3}$ | | Yaws (2003) | X | 237 |
| | $1.6\times10^{-3}$ | | Duchowicz et al. (2020) | Q | |
| | $1.6\times10^{-3}$ | | Duchowicz et al. (2020) | Q | |
| | $1.8\times10^{-3}$ | | Wang et al. (2017) | Q | 80, 238 |
| | $1.1\times10^{-3}$ | | Wang et al. (2017) | Q | 80, 239 |
| | $2.5\times10^{-3}$ | | Wang et al. (2017) | Q | 80, 240 |
| | $1.5\times10^{-3}$ | | Gharagheizi et al. (2010) | Q | 246 |
| | $1.4\times10^{-3}$ | | Hilal et al. (2008) | Q | |
| | $7.1\times10^{-4}$ | | Modarresi et al. (2007) | Q | 67 |
| | $2.1\times10^{-3}$ | | Yaffe et al. (2003) | Q | 248, 249 |
| | $1.7\times10^{-3}$ | | English and Carroll (2001) | Q | 230, 260 |
| | $5.1\times10^{-4}$ | | Katritzky et al. (1998) | Q | |
| | $9.5\times10^{-4}$ | | Nirmalakhandan et al. (1997) | Q | |
| | $2.0\times10^{-3}$ | | Yaws (1999) | ? | 21 |
| | $2.0\times10^{-3}$ | | Yaws and Yang (1992) | ? | 21 |





Table A2.5: Mononuclear aromatics (. . . continued)

| Substance Formula (Trivial Name) [CAS Registry Number] InChIKey | $H_s^{cp}$ (at $T^\ominus$) $\left[\dfrac{\mathrm{mol}}{\mathrm{m^3\,Pa}}\right]$ | $\dfrac{\mathrm{d}\ln H_s^{cp}}{\mathrm{d}(1/T)}$ [K] | Reference | Type | Note |
|---|---|---|---|---|---|
| butylbenzene | $7.1\times10^{-4}$ | 5100 | Brockbank (2013) | L | 1 |
| $C_6H_5C_4H_9$ | $7.1\times10^{-4}$ | 5500 | Plyasunov and Shock (2000) | L | |
| [104-51-8] | $7.7\times10^{-4}$ | | Mackay and Shiu (1981) | L | |
| OCKPCBLVNKHBMX-UHFFFAOYSA-N | $2.0\times10^{-3}$ | 4500 | Hiatt (2013) | M | |
| | $7.4\times10^{-4}$ | | Ryu and Park (1999) | M | |
| | $9.1\times10^{-4}$ | 2700 | Kondoh and Nakajima (1997) | M | |
| | $6.2\times10^{-4}$ | 6000 | Perlinger et al. (1993) | M | |
| | $7.1\times10^{-4}$ | | Li and Carr (1993) | M | |
| | $6.7\times10^{-4}$ | | Li et al. (1993) | M | |
| | $6.2\times10^{-4}$ | | Duchowicz et al. (2020) | V | 186 |
| | $6.2\times10^{-4}$ | | HSDB (2015) | V | |
| | $9.9\times10^{-5}$ | | Abraham and Acree (2007) | V | |
| | $7.5\times10^{-4}$ | | Mackay et al. (2006a) | V | |
| | $7.5\times10^{-4}$ | | Shiu and Ma (2000) | V | |
| | $7.5\times10^{-4}$ | | Mackay et al. (1992a) | V | |
| | $7.6\times10^{-4}$ | | Meylan and Howard (1991) | V | |
| | $7.5\times10^{-4}$ | | Eastcott et al. (1988) | V | |
| | $7.4\times10^{-4}$ | | Abraham (1984) | V | |
| | $1.7\times10^{-3}$ | 6500 | Ben-Naim and Wilf (1980) | V | 1 |
| | $7.9\times10^{-4}$ | | Hine and Mookerjee (1975) | V | |
| | $7.2\times10^{-4}$ | 5300 | Owens et al. (1986) | T | |
| | $7.5\times10^{-4}$ | | Yaws (2003) | X | 237 |
| | $3.1\times10^{-3}$ | | Duchowicz et al. (2020) | Q | |
| | $7.6\times10^{-4}$ | | Gharagheizi et al. (2012) | Q | |
| | $6.2\times10^{-4}$ | | Raventos-Duran et al. (2010) | Q | 271, 243 |
| | $4.9\times10^{-4}$ | | Raventos-Duran et al. (2010) | Q | 244 |
| | $7.8\times10^{-4}$ | | Raventos-Duran et al. (2010) | Q | 245 |
| | $7.1\times10^{-4}$ | | Gharagheizi et al. (2010) | Q | 246 |
| | $7.7\times10^{-4}$ | | Hilal et al. (2008) | Q | |
| | $4.9\times10^{-4}$ | | Modarresi et al. (2007) | Q | 67 |
| | | 5300 | Kühne et al. (2005) | Q | |
| | $7.2\times10^{-4}$ | | Yaffe et al. (2003) | Q | 248, 272 |
| | $6.4\times10^{-4}$ | | Yao et al. (2002) | Q | 229 |
| | $7.9\times10^{-4}$ | | English and Carroll (2001) | Q | 230, 231 |
| | $5.7\times10^{-4}$ | | Katritzky et al. (1998) | Q | |
| | $6.9\times10^{-4}$ | | Russell et al. (1992) | Q | 358 |
| | $6.9\times10^{-4}$ | | Suzuki et al. (1992) | Q | 232 |
| | $7.1\times10^{-4}$ | | Meylan and Howard (1991) | Q | |
| | $8.4\times10^{-4}$ | | Nirmalakhandan and Speece (1988) | Q | |
| | | 4900 | Kühne et al. (2005) | ? | |
| | $7.6\times10^{-4}$ | | Yaws (1999) | ? | 21 |
| | $7.5\times10^{-4}$ | | Yaws and Yang (1992) | ? | 21 |
| | $7.5\times10^{-4}$ | | Abraham et al. (1990) | ? | |





Table A2.5: Mononuclear aromatics (. . . continued)

| Substance Formula (Trivial Name) [CAS Registry Number] InChIKey | $H_s^{cp}$ (at $T^{\ominus}$) $\left[\dfrac{\text{mol}}{\text{m}^3\,\text{Pa}}\right]$ | $\dfrac{\text{d}\ln H_s^{cp}}{\text{d}(1/T)}$ [K] | Reference | Type | Note |
|---|---|---|---|---|---|
| (1-methylpropyl)-benzene | $5.3\times10^{-4}$ | | Plyasunov and Shock (2000) | L | |
| $C_6H_5C_4H_9$ | $7.1\times10^{-4}$ | | Mackay and Shiu (1981) | L | |
| (*sec*-butylbenzene) | $1.3\times10^{-3}$ | 4600 | Hiatt (2013) | M | |
| [135-98-8] | $7.5\times10^{-4}$ | 2300 | Kondoh and Nakajima (1997) | M | |
| ZJMWRROPUADPEA-UHFFFAOYSA-N | $5.6\times10^{-4}$ | | Duchowicz et al. (2020) | V | 186 |
| | $5.5\times10^{-4}$ | | HSDB (2015) | V | |
| | $5.3\times10^{-4}$ | | Mackay et al. (2006a) | V | |
| | $5.3\times10^{-4}$ | | Mackay et al. (1992a) | V | |
| | $5.4\times10^{-4}$ | | Eastcott et al. (1988) | V | |
| | $8.6\times10^{-4}$ | | Hine and Mookerjee (1975) | V | |
| | $5.9\times10^{-4}$ | | Yaws (2003) | X | 237 |
| | $1.2\times10^{-3}$ | | Duchowicz et al. (2020) | Q | |
| | $5.9\times10^{-4}$ | | Gharagheizi et al. (2012) | Q | |
| | $6.2\times10^{-4}$ | | Raventos-Duran et al. (2010) | Q | 242, 243 |
| | $7.8\times10^{-4}$ | | Raventos-Duran et al. (2010) | Q | 244 |
| | $7.8\times10^{-4}$ | | Raventos-Duran et al. (2010) | Q | 245 |
| | $6.9\times10^{-4}$ | | Gharagheizi et al. (2010) | Q | 246 |
| | $8.6\times10^{-4}$ | | Hilal et al. (2008) | Q | |
| | $6.9\times10^{-4}$ | | Modarresi et al. (2007) | Q | 67 |
| | $7.2\times10^{-4}$ | | Yaffe et al. (2003) | Q | 248, 249 |
| | $9.0\times10^{-4}$ | | English and Carroll (2001) | Q | 230, 231 |
| | $9.9\times10^{-5}$ | | Nirmalakhandan et al. (1997) | Q | |
| | $6.1\times10^{-4}$ | | Suzuki et al. (1992) | Q | 232 |
| | $7.2\times10^{-4}$ | | Nirmalakhandan and Speece (1988) | Q | |
| | $1.3\times10^{-3}$ | | Yaws (1999) | ? | 21 |
| (2-methylpropyl)-benzene | $2.9\times10^{-4}$ | | Plyasunov and Shock (2000) | L | |
| $C_6H_5C_4H_9$ | $3.0\times10^{-4}$ | | Mackay and Shiu (1981) | L | |
| (isobutylbenzene) | $2.9\times10^{-4}$ | | Duchowicz et al. (2020) | V | 186 |
| [538-93-2] | $3.0\times10^{-4}$ | | Mackay et al. (2006a) | V | |
| KXUHSQYYJYAXGZ-UHFFFAOYSA-N | $3.0\times10^{-4}$ | | Mackay et al. (1992a) | V | |
| | $3.0\times10^{-4}$ | | Eastcott et al. (1988) | V | |
| | $5.5\times10^{-4}$ | | Yaws (2003) | X | 237 |
| | $1.2\times10^{-3}$ | | Duchowicz et al. (2020) | Q | |
| | $6.2\times10^{-4}$ | | Gharagheizi et al. (2012) | Q | |
| | $6.9\times10^{-4}$ | | Gharagheizi et al. (2010) | Q | 246 |
| | $7.0\times10^{-4}$ | | Hilal et al. (2008) | Q | |
| | $5.2\times10^{-4}$ | | Modarresi et al. (2007) | Q | 67 |
| | $3.1\times10^{-4}$ | | Yaffe et al. (2003) | Q | 248, 249 |
| | $4.3\times10^{-4}$ | | Yao et al. (2002) | Q | 229 |
| | $7.7\times10^{-4}$ | | English and Carroll (2001) | Q | 230, 231 |
| | $5.4\times10^{-4}$ | | Katritzky et al. (1998) | Q | |
| | $7.0\times10^{-4}$ | | Nirmalakhandan et al. (1997) | Q | |
| | $1.3\times10^{-3}$ | | Yaws (1999) | ? | 21 |



Table A2.5: Mononuclear aromatics (...continued)

| Substance Formula (Trivial Name) [CAS Registry Number] InChIKey | $H_s^{cp}$ (at $T^\ominus$) $\left[\dfrac{\text{mol}}{\text{m}^3\,\text{Pa}}\right]$ | $\dfrac{\text{d}\ln H_s^{cp}}{\text{d}(1/T)}$ [K] | Reference | Type | Note |
|---|---|---|---|---|---|
| (1,1-dimethylethyl)-benzene | $7.9\times10^{-4}$ | | Plyasunov and Shock (2000) | L | |
| $C_6H_5C_4H_9$ | $8.3\times10^{-4}$ | | Mackay and Shiu (1981) | L | |
| (*tert*-butylbenzene) | $1.6\times10^{-3}$ | 4700 | Hiatt (2013) | M | |
| [98-06-6] | $9.4\times10^{-4}$ | 2400 | Kondoh and Nakajima (1997) | M | |
| YTZKOQUCBOVLHL-UHFFFAOYSA-N | $7.5\times10^{-4}$ | | Duchowicz et al. (2020) | V | 186 |
| | $7.5\times10^{-4}$ | | HSDB (2015) | V | |
| | $7.8\times10^{-4}$ | | Mackay et al. (2006a) | V | |
| | $7.8\times10^{-4}$ | | Mackay et al. (1992a) | V | |
| | $7.7\times10^{-4}$ | | Eastcott et al. (1988) | V | |
| | $8.4\times10^{-4}$ | | Hine and Mookerjee (1975) | V | |
| | $6.1\times10^{-4}$ | | Yaws (2003) | X | 237 |
| | $5.3\times10^{-4}$ | | Duchowicz et al. (2020) | Q | |
| | $5.3\times10^{-4}$ | | Gharagheizi et al. (2012) | Q | |
| | $5.6\times10^{-4}$ | | Gharagheizi et al. (2010) | Q | 246 |
| | $7.7\times10^{-4}$ | | Hilal et al. (2008) | Q | |
| | $6.6\times10^{-4}$ | | Modarresi et al. (2007) | Q | 67 |
| | $8.6\times10^{-4}$ | | Yaffe et al. (2003) | Q | 248, 249 |
| | $9.0\times10^{-4}$ | | English and Carroll (2001) | Q | 230, 231 |
| | $4.8\times10^{-4}$ | | Katritzky et al. (1998) | Q | |
| | $5.3\times10^{-4}$ | | Suzuki et al. (1992) | Q | 232 |
| | $6.0\times10^{-4}$ | | Nirmalakhandan and Speece (1988) | Q | |
| | $1.4\times10^{-3}$ | | Yaws (1999) | ? | 21 |
| 1-methyl-2-propylbenzene | $7.2\times10^{-4}$ | | Yaws (2003) | X | 237 |
| $C_{10}H_{14}$ | $5.3\times10^{-4}$ | | Gharagheizi et al. (2012) | Q | |
| [1074-17-5] | $1.1\times10^{-3}$ | | Gharagheizi et al. (2010) | Q | 246 |
| YQZBFMJOASEONC-UHFFFAOYSA-N | | | | | |
| 1-methyl-3-propylbenzene | $7.7\times10^{-4}$ | | Yaws (2003) | X | 237 |
| $C_{10}H_{14}$ | $4.9\times10^{-4}$ | | Gharagheizi et al. (2012) | Q | |
| [1074-43-7] | $1.1\times10^{-3}$ | | Gharagheizi et al. (2010) | Q | 246 |
| QUEBYVKXYIKVSO-UHFFFAOYSA-N | | | | | |
| 1-methyl-4-propylbenzene | $9.6\times10^{-4}$ | | Yaws (2003) | X | 237 |
| $C_{10}H_{14}$ | $4.7\times10^{-4}$ | | Gharagheizi et al. (2012) | Q | |
| [1074-55-1] | $1.1\times10^{-3}$ | | Gharagheizi et al. (2010) | Q | 246 |
| JXFVMNFKABWTHD-UHFFFAOYSA-N | | | | | |
| 2-ethyl-*m*-xylene | $1.3\times10^{-3}$ | | Yaws (2003) | X | 237 |
| $C_{10}H_{14}$ | $4.6\times10^{-4}$ | | Gharagheizi et al. (2012) | Q | |
| [2870-04-4] | $1.1\times10^{-3}$ | | Gharagheizi et al. (2010) | Q | 246 |
| CHIKRULMSSADAF-UHFFFAOYSA-N | $1.3\times10^{-3}$ | | Yaws (1999) | ? | 21 |
| 2-ethyl-*p*-xylene | $1.2\times10^{-3}$ | | Yaws (2003) | X | 237 |
| $C_{10}H_{14}$ | $3.9\times10^{-4}$ | | Gharagheizi et al. (2012) | Q | |
| [1758-88-9] | $1.1\times10^{-3}$ | | Gharagheizi et al. (2010) | Q | 246 |
| AXIUBBVSOWPLDA-UHFFFAOYSA-N | $1.2\times10^{-3}$ | | Yaws (1999) | ? | 21 |





Table A2.5: Mononuclear aromatics (...continued)

| Substance Formula (Trivial Name) [CAS Registry Number] InChIKey | $H_s^{cp}$ (at $T^{\ominus}$) $\left[\dfrac{\text{mol}}{\text{m}^3\,\text{Pa}}\right]$ | $\dfrac{\text{d}\ln H_s^{cp}}{\text{d}(1/T)}$ [K] | Reference | Type | Note |
|---|---|---|---|---|---|
| 3-ethyl-$o$-xylene | $1.2\times10^{-3}$ | | Yaws (2003) | X | 237 |
| C$_{10}$H$_{14}$ | $4.7\times10^{-4}$ | | Gharagheizi et al. (2012) | Q | |
| [933-98-2] | $1.1\times10^{-3}$ | | Gharagheizi et al. (2010) | Q | 246 |
| QUBBAXISAHIDNM-UHFFFAOYSA-N | $1.2\times10^{-3}$ | | Yaws (1999) | ? | 21 |
| 4-ethyl-$o$-xylene | $1.3\times10^{-3}$ | | Yaws (2003) | X | 237 |
| C$_{10}$H$_{14}$ | $4.2\times10^{-4}$ | | Gharagheizi et al. (2012) | Q | |
| [934-80-5] | $1.1\times10^{-3}$ | | Gharagheizi et al. (2010) | Q | 246 |
| SBUYFICWQNHBCM-UHFFFAOYSA-N | $4.5\times10^{-4}$ | | Yao et al. (2002) | Q | 229 |
| | $1.3\times10^{-3}$ | | Yaws (1999) | ? | 21 |
| 4-ethyl-$m$-xylene | $1.1\times10^{-3}$ | | Yaws (2003) | X | 237 |
| C$_{10}$H$_{14}$ | $4.0\times10^{-4}$ | | Gharagheizi et al. (2012) | Q | |
| [874-41-9] | $1.1\times10^{-3}$ | | Gharagheizi et al. (2010) | Q | 246 |
| MEMBJMDZWKVOTB-UHFFFAOYSA-N | $3.0\times10^{-4}$ | | Yao et al. (2002) | Q | 229 |
| | $1.1\times10^{-3}$ | | Yaws (1999) | ? | 21 |
| 5-ethyl-$m$-xylene | $1.3\times10^{-3}$ | | Yaws (2003) | X | 237 |
| C$_{10}$H$_{14}$ | $1.1\times10^{-3}$ | | Wang et al. (2017) | Q | 80, 238 |
| (1-ethyl-3,5-dimethylbenzene) | $9.3\times10^{-4}$ | | Wang et al. (2017) | Q | 80, 239 |
| [934-74-7] | $2.2\times10^{-3}$ | | Wang et al. (2017) | Q | 80, 240 |
| LMAUULKNZLEMGN-UHFFFAOYSA-N | $3.3\times10^{-4}$ | | Gharagheizi et al. (2012) | Q | |
| | $1.1\times10^{-3}$ | | Gharagheizi et al. (2010) | Q | 246 |
| | $1.3\times10^{-3}$ | | Yaws (1999) | ? | 21 |
| 1-methyl-2-(1-methylethyl)-benzene | $1.6\times10^{-3}$ | | Plyasunov and Shock (2000) | L | |
| C$_{10}$H$_{14}$ | $8.7\times10^{-4}$ | | Duchowicz et al. (2020) | V | 186 |
| ($o$-cymene) | $9.0\times10^{-4}$ | | HSDB (2015) | V | |
| [527-84-4] | $1.2\times10^{-3}$ | | Yaws (2003) | X | 258 |
| WWRCMNKATXZARA-UHFFFAOYSA-N | $1.2\times10^{-3}$ | | Yaws (2003) | X | 237 |
| | $1.3\times10^{-3}$ | | Dupeux et al. (2022) | Q | 259 |
| | $6.1\times10^{-4}$ | | Duchowicz et al. (2020) | Q | |
| | $4.6\times10^{-4}$ | | Gharagheizi et al. (2012) | Q | |
| | $6.2\times10^{-4}$ | | Raventos-Duran et al. (2010) | Q | 271, 243 |
| | $9.9\times10^{-4}$ | | Raventos-Duran et al. (2010) | Q | 244 |
| | $7.8\times10^{-4}$ | | Raventos-Duran et al. (2010) | Q | 245 |
| | $1.2\times10^{-3}$ | | Gharagheizi et al. (2010) | Q | 246 |
| | $1.2\times10^{-3}$ | | Hilal et al. (2008) | Q | |
| | $6.6\times10^{-4}$ | | Modarresi et al. (2007) | Q | 67 |
| | $5.3\times10^{-4}$ | | Katritzky et al. (1998) | Q | |
| | $1.2\times10^{-3}$ | | Yaws (1999) | ? | 21 |



Table A2.5: Mononuclear aromatics (…continued)

| Substance Formula (Trivial Name) [CAS Registry Number] InChIKey | $H_s^{cp}$ (at $T^\ominus$) $\left[\dfrac{\mathrm{mol}}{\mathrm{m^3\,Pa}}\right]$ | $\dfrac{\mathrm{d}\ln H_s^{cp}}{\mathrm{d}(1/T)}$ [K] | Reference | Type | Note |
|---|---|---|---|---|---|
| 1-methyl-3-(1-methylethyl)-benzene | $1.4\times10^{-3}$ | | Plyasunov and Shock (2000) | L | |
| $C_{10}H_{14}$ | $1.4\times10^{-3}$ | | Duchowicz et al. (2020) | V | 186 |
| ($m$-cymene) | $1.4\times10^{-3}$ | | HSDB (2015) | V | |
| [535-77-3] | $9.0\times10^{-4}$ | | Copolovici and Niinemets (2005) | V | |
| XCYJPXQACVEIOS-UHFFFAOYSA-N | $1.2\times10^{-3}$ | | Yaws (2003) | X | 258 |
| | $1.2\times10^{-3}$ | | Yaws (2003) | X | 237 |
| | $8.2\times10^{-4}$ | | Dupeux et al. (2022) | Q | 259 |
| | $6.1\times10^{-4}$ | | Duchowicz et al. (2020) | Q | |
| | $4.2\times10^{-4}$ | | Gharagheizi et al. (2012) | Q | |
| | $1.2\times10^{-3}$ | | Gharagheizi et al. (2010) | Q | 246 |
| | $8.6\times10^{-4}$ | | Hilal et al. (2008) | Q | |
| | $4.5\times10^{-4}$ | | Modarresi et al. (2007) | Q | 67 |
| | $1.2\times10^{-3}$ | | Yaws (1999) | ? | 21 |
| 1-methyl-4-(1-methylethyl)-benzene | $9.6\times10^{-4}$ | | Plyasunov and Shock (2000) | L | |
| $C_{10}H_{14}$ | $1.3\times10^{-3}$ | | Mackay and Shiu (1981) | L | |
| ($p$-cymene; $p$-isopropyltoluene) | $1.8\times10^{-3}$ | 4900 | Hiatt (2013) | M | |
| [99-87-6] | $1.0\times10^{-3}$ | 2600 | Kondoh and Nakajima (1997) | M | |
| HFPZCAJZSCWRBC-UHFFFAOYSA-N | $9.0\times10^{-4}$ | | Duchowicz et al. (2020) | V | 186 |
| | $8.0\times10^{-4}$ | | Duchowicz et al. (2020) | V | 186 |
| | $1.1\times10^{-3}$ | | Martins et al. (2017) | V | 315 |
| | $9.0\times10^{-4}$ | | HSDB (2015) | V | |
| | $1.2\times10^{-3}$ | | Mackay et al. (2006a) | V | |
| | $1.1\times10^{-3}$ | | Copolovici and Niinemets (2005) | V | |
| | $9.1\times10^{-4}$ | | Niinemets and Reichstein (2002) | V | |
| | $1.3\times10^{-3}$ | | Abraham et al. (1994a) | V | |
| | $1.2\times10^{-3}$ | | Mackay et al. (1992a) | V | |
| | $1.2\times10^{-3}$ | | Eastcott et al. (1988) | V | |
| | $1.3\times10^{-3}$ | | Yaws (2003) | X | 258 |
| | $1.3\times10^{-3}$ | | Yaws (2003) | X | 237 |
| | $7.6\times10^{-4}$ | | Dupeux et al. (2022) | Q | 259 |
| | $6.1\times10^{-4}$ | | Duchowicz et al. (2020) | Q | |
| | $6.1\times10^{-4}$ | | Duchowicz et al. (2020) | Q | |
| | $4.2\times10^{-4}$ | | Gharagheizi et al. (2012) | Q | |
| | $6.2\times10^{-4}$ | | Raventos-Duran et al. (2010) | Q | 271, 243 |
| | $7.8\times10^{-4}$ | | Raventos-Duran et al. (2010) | Q | 244 |
| | $7.8\times10^{-4}$ | | Raventos-Duran et al. (2010) | Q | 245 |
| | $1.2\times10^{-3}$ | | Gharagheizi et al. (2010) | Q | 246 |
| | $8.8\times10^{-4}$ | | Hilal et al. (2008) | Q | |
| | $5.4\times10^{-4}$ | | Modarresi et al. (2007) | Q | 67 |
| | $5.5\times10^{-4}$ | | Modarresi et al. (2007) | Q | 67 |
| | | 5300 | Kühne et al. (2005) | Q | |
| | $8.0\times10^{-4}$ | | Yaffe et al. (2003) | Q | 248, 249 |
| | $1.4\times10^{-3}$ | | English and Carroll (2001) | Q | 230, 274 |
| | $5.4\times10^{-4}$ | | Katritzky et al. (1998) | Q | |
| | $6.5\times10^{-4}$ | | Nirmalakhandan et al. (1997) | Q | |





Table A2.5: Mononuclear aromatics (...continued)

| Substance Formula (Trivial Name) [CAS Registry Number] InChIKey | $H_s^{cp}$ (at $T^\ominus$) $\left[\dfrac{\text{mol}}{\text{m}^3\,\text{Pa}}\right]$ | $\dfrac{\text{d}\ln H_s^{cp}}{\text{d}(1/T)}$ [K] | Reference | Type | Note |
|---|---|---|---|---|---|
| | | 4500 | Kühne et al. (2005) | ? | |
| | $1.3\times10^{-3}$ | | Yaws (1999) | ? | 21 |
| 4-*tert*-butyltoluene | $7.8\times10^{-4}$ | | Yaws (2003) | X | 237 |
| $C_{11}H_{16}$ | $6.4\times10^{-4}$ | | HSDB (2015) | Q | 99 |
| [98-51-1] | $2.7\times10^{-4}$ | | Gharagheizi et al. (2012) | Q | |
| QCWXDVFBZVHKLV-UHFFFAOYSA-N | $6.4\times10^{-4}$ | | Zhang et al. (2010) | Q | 287, 288 |
| | $5.2\times10^{-4}$ | | Zhang et al. (2010) | Q | 287, 289 |
| | $1.3\times10^{-3}$ | | Zhang et al. (2010) | Q | 287, 290 |
| | $4.7\times10^{-4}$ | | Zhang et al. (2010) | Q | 287, 291 |
| | $8.2\times10^{-4}$ | | Gharagheizi et al. (2010) | Q | 246 |
| pentylbenzene | $6.0\times10^{-4}$ | 6000 | Brockbank (2013) | L | 1, 359 |
| $C_6H_5C_5H_{11}$ | $5.1\times10^{-4}$ | 5900 | Plyasunov and Shock (2000) | L | |
| [538-68-1] | $1.7\times10^{-3}$ | | Mackay and Shiu (1981) | L | |
| PWATWSYOIIXYMA-UHFFFAOYSA-N | $6.1\times10^{-4}$ | | Ryu and Park (1999) | M | |
| | $3.9\times10^{-4}$ | | Duchowicz et al. (2020) | V | 186 |
| | $5.9\times10^{-4}$ | | Mackay et al. (2006a) | V | |
| | $5.9\times10^{-4}$ | | Mackay et al. (1992a) | V | |
| | $1.6\times10^{-3}$ | | Eastcott et al. (1988) | V | |
| | $6.0\times10^{-4}$ | | Abraham (1984) | V | |
| | $3.0\times10^{-3}$ | 7800 | Ben-Naim and Wilf (1980) | V | 1 |
| | $5.3\times10^{-4}$ | 5900 | Owens et al. (1986) | T | |
| | $4.4\times10^{-4}$ | | Yaws (2003) | X | 237 |
| | $3.1\times10^{-3}$ | | Duchowicz et al. (2020) | Q | |
| | $5.6\times10^{-4}$ | | Gharagheizi et al. (2012) | Q | |
| | $4.9\times10^{-4}$ | | Raventos-Duran et al. (2010) | Q | 271, 243 |
| | $3.9\times10^{-4}$ | | Raventos-Duran et al. (2010) | Q | 244 |
| | $4.9\times10^{-4}$ | | Raventos-Duran et al. (2010) | Q | 245 |
| | $6.0\times10^{-4}$ | | Gharagheizi et al. (2010) | Q | 246 |
| | $6.1\times10^{-4}$ | | Hilal et al. (2008) | Q | |
| | $4.3\times10^{-4}$ | | Modarresi et al. (2007) | Q | 67 |
| | $6.1\times10^{-4}$ | | Yaffe et al. (2003) | Q | 248, 249 |
| | $4.4\times10^{-4}$ | | Yao et al. (2002) | Q | 229 |
| | $5.8\times10^{-4}$ | | English and Carroll (2001) | Q | 230, 260 |
| | $5.7\times10^{-4}$ | | Katritzky et al. (1998) | Q | |
| | $6.4\times10^{-4}$ | | Nirmalakhandan et al. (1997) | Q | |
| | $4.4\times10^{-4}$ | | Yaws (1999) | ? | 21 |
| | $5.9\times10^{-4}$ | | Yaws and Yang (1992) | ? | 21 |
| | $6.0\times10^{-4}$ | | Abraham et al. (1990) | ? | |
| pentamethylbenzene | $7.7\times10^{-3}$ | | Hilal et al. (2008) | Q | |
| $C_{11}H_{16}$ | $7.0\times10^{-4}$ | | Modarresi et al. (2007) | Q | 67 |
| [700-12-9] | | | | | |
| BEZDDPMMPIDMGJ-UHFFFAOYSA-N | | | | | |





Table A2.5: Mononuclear aromatics (...continued)

| Substance Formula (Trivial Name) [CAS Registry Number] InChIKey | $H_s^{cp}$ (at $T^{\ominus}$) $\left[\dfrac{\text{mol}}{\text{m}^3\,\text{Pa}}\right]$ | $\dfrac{\text{d}\ln H_s^{cp}}{\text{d}(1/T)}$ [K] | Reference | Type | Note |
|---|---|---|---|---|---|
| 1,2-dimethyl-3-propylbenzene $C_{11}H_{16}$ [17059-44-8] IRUSTUOJENXLMN-UHFFFAOYSA-N | $8.7\times10^{-4}$ $2.9\times10^{-4}$ $7.9\times10^{-4}$ | | Yaws (2003) Gharagheizi et al. (2012) Gharagheizi et al. (2010) | X Q Q | 237 246 |
| 1,2-dimethyl-4-propylbenzene $C_{11}H_{16}$ [3982-66-9] FZJVYOOQGFZCSY-UHFFFAOYSA-N | $8.8\times10^{-4}$ $2.8\times10^{-4}$ $7.9\times10^{-4}$ | | Yaws (2003) Gharagheizi et al. (2012) Gharagheizi et al. (2010) | X Q Q | 237 246 |
| 1,3-dimethyl-2-propylbenzene $C_{11}H_{16}$ [17059-45-9] POVRSTNZQPBWAS-UHFFFAOYSA-N | $8.9\times10^{-4}$ $2.7\times10^{-4}$ $7.9\times10^{-4}$ | | Yaws (2003) Gharagheizi et al. (2012) Gharagheizi et al. (2010) | X Q Q | 237 246 |
| 1,3-dimethyl-4-propylbenzene $C_{11}H_{16}$ [61827-85-8] HPAXKQMKDWCLGU-UHFFFAOYSA-N | $9.0\times10^{-4}$ $2.6\times10^{-4}$ $7.9\times10^{-4}$ | | Yaws (2003) Gharagheizi et al. (2012) Gharagheizi et al. (2010) | X Q Q | 237 246 |
| 1,3-dimethyl-5-propylbenzene $C_{11}H_{16}$ [3982-64-7] NBICXWXPZRNMPF-UHFFFAOYSA-N | $9.3\times10^{-4}$ $2.3\times10^{-4}$ $7.9\times10^{-4}$ | | Yaws (2003) Gharagheizi et al. (2012) Gharagheizi et al. (2010) | X Q Q | 237 246 |
| 1,4-dimethyl-2-propylbenzene $C_{11}H_{16}$ [3042-50-0] PWEDYOIWLPZSRP-UHFFFAOYSA-N | $9.2\times10^{-4}$ $2.4\times10^{-4}$ $7.9\times10^{-4}$ | | Yaws (2003) Gharagheizi et al. (2012) Gharagheizi et al. (2010) | X Q Q | 237 246 |
| 1,2-dimethyl-3-isopropylbenzene $C_{11}H_{16}$ [22539-65-7] GDEQPEBFOWYWSA-UHFFFAOYSA-N | $8.4\times10^{-4}$ $2.4\times10^{-4}$ $9.4\times10^{-4}$ | | Yaws (2003) Gharagheizi et al. (2012) Gharagheizi et al. (2010) | X Q Q | 237 246 |
| 1,2-dimethyl-4-isopropylbenzene $C_{11}H_{16}$ [4132-77-8] MGMSKQZIAGFMRU-UHFFFAOYSA-N | $8.5\times10^{-4}$ $2.3\times10^{-4}$ $9.4\times10^{-4}$ | | Yaws (2003) Gharagheizi et al. (2012) Gharagheizi et al. (2010) | X Q Q | 237 246 |
| 1,3-dimethyl-2-isopropylbenzene $C_{11}H_{16}$ [14411-75-7] IVCIQLTVLDOHKA-UHFFFAOYSA-N | $8.7\times10^{-4}$ $2.1\times10^{-4}$ $9.4\times10^{-4}$ | | Yaws (2003) Gharagheizi et al. (2012) Gharagheizi et al. (2010) | X Q Q | 237 246 |
| 1,3-dimethyl-4-isopropylbenzene $C_{11}H_{16}$ [4706-89-2] AADQFNAACHHRLT-UHFFFAOYSA-N | $8.7\times10^{-4}$ $2.1\times10^{-4}$ $9.4\times10^{-4}$ | | Yaws (2003) Gharagheizi et al. (2012) Gharagheizi et al. (2010) | X Q Q | 237 246 |



Table A2.5: Mononuclear aromatics (...continued)

| Substance Formula (Trivial Name) [CAS Registry Number] InChIKey | $H_s^{cp}$ (at $T^\ominus$) $\left[\dfrac{\text{mol}}{\text{m}^3\,\text{Pa}}\right]$ | $\dfrac{\text{d}\ln H_s^{cp}}{\text{d}(1/T)}$ [K] | Reference | Type | Note |
|---|---|---|---|---|---|
| 1,3-dimethyl-5-isopropylbenzene $C_{11}H_{16}$ [4706-90-5] RMKJTYPFCFNTGQ-UHFFFAOYSA-N | $9.2\times10^{-4}$ $1.9\times10^{-4}$ $9.4\times10^{-4}$ | | Yaws (2003) Gharagheizi et al. (2012) Gharagheizi et al. (2010) | X Q Q | 237 246 |
| 1,4-dimethyl-2-isopropylbenzene $C_{11}H_{16}$ [4132-72-3] CLSBTDGUHSQYTO-UHFFFAOYSA-N | $9.0\times10^{-4}$ $1.9\times10^{-4}$ $9.4\times10^{-4}$ | | Yaws (2003) Gharagheizi et al. (2012) Gharagheizi et al. (2010) | X Q Q | 237 246 |
| 1-ethyl-2-propylbenzene $C_{11}H_{16}$ [16021-20-8] DMUVQFCRCMDZPW-UHFFFAOYSA-N | $9.4\times10^{-4}$ $3.5\times10^{-4}$ $6.2\times10^{-4}$ $6.2\times10^{-4}$ $6.2\times10^{-4}$ $8.8\times10^{-4}$ | | Yaws (2003) Gharagheizi et al. (2012) Raventos-Duran et al. (2010) Raventos-Duran et al. (2010) Raventos-Duran et al. (2010) Gharagheizi et al. (2010) | X Q Q Q Q Q | 237 242, 243 244 245 246 |
| 1-ethyl-3-propylbenzene $C_{11}H_{16}$ [20024-91-3] QCYGXOCMWHSXSU-UHFFFAOYSA-N | $9.6\times10^{-4}$ $3.3\times10^{-4}$ $8.8\times10^{-4}$ | | Yaws (2003) Gharagheizi et al. (2012) Gharagheizi et al. (2010) | X Q Q | 237 246 |
| 1-ethyl-4-propylbenzene $C_{11}H_{16}$ [20024-90-2] ADQDTIAWIXUACV-UHFFFAOYSA-N | $9.2\times10^{-4}$ $3.7\times10^{-4}$ $8.8\times10^{-4}$ | | Yaws (2003) Gharagheizi et al. (2012) Gharagheizi et al. (2010) | X Q Q | 237 246 |
| 1-ethyl-2-isopropylbenzene $C_{11}H_{16}$ [18970-44-0] ZAJYARZMPOEGLK-UHFFFAOYSA-N | $9.8\times10^{-4}$ $2.5\times10^{-4}$ $9.1\times10^{-4}$ | | Yaws (2003) Gharagheizi et al. (2012) Gharagheizi et al. (2010) | X Q Q | 237 246 |
| 1-ethyl-3-isopropylbenzene $C_{11}H_{16}$ [4920-99-4] GSLSBTNLESMZTN-UHFFFAOYSA-N | $9.6\times10^{-4}$ $2.6\times10^{-4}$ $9.1\times10^{-4}$ | | Yaws (2003) Gharagheizi et al. (2012) Gharagheizi et al. (2010) | X Q Q | 237 246 |
| 1-ethyl-4-isopropylbenzene $C_{11}H_{16}$ [4218-48-8] GUUDUUDWUWUTPD-UHFFFAOYSA-N | $9.1\times10^{-4}$ $3.0\times10^{-4}$ $9.1\times10^{-4}$ | | Yaws (2003) Gharagheizi et al. (2012) Gharagheizi et al. (2010) | X Q Q | 237 246 |
| 1-methyl-2,3-diethylbenzene $C_{11}H_{16}$ [13632-93-4] LRJOXARIJKBUFE-UHFFFAOYSA-N | $9.0\times10^{-4}$ $2.6\times10^{-4}$ $7.9\times10^{-4}$ | | Yaws (2003) Gharagheizi et al. (2012) Gharagheizi et al. (2010) | X Q Q | 237 246 |
| 1-methyl-2,4-diethylbenzene $C_{11}H_{16}$ [1758-85-6] PZMJNJDRDKPVLB-UHFFFAOYSA-N | $9.1\times10^{-4}$ $2.5\times10^{-4}$ $7.9\times10^{-4}$ | | Yaws (2003) Gharagheizi et al. (2012) Gharagheizi et al. (2010) | X Q Q | 237 246 |



Table A2.5: Mononuclear aromatics (...continued)

| Substance Formula (Trivial Name) [CAS Registry Number] InChIKey | $H_s^{cp}$ (at $T^{\ominus}$) $\left[\dfrac{\mathrm{mol}}{\mathrm{m^3\,Pa}}\right]$ | $\dfrac{\mathrm{d}\ln H_s^{cp}}{\mathrm{d}(1/T)}$ [K] | Reference | Type | Note |
|---|---|---|---|---|---|
| 1-methyl-2,5-diethylbenzene $C_{11}H_{16}$ [13632-94-5] ZEHGGUIGEDITMM-UHFFFAOYSA-N | $9.0{\times}10^{-4}$ $2.6{\times}10^{-4}$ $7.9{\times}10^{-4}$ | | Yaws (2003) Gharagheizi et al. (2012) Gharagheizi et al. (2010) | X Q Q | 237 246 |
| 1-methyl-2,6-diethylbenzene $C_{11}H_{16}$ [13632-95-6] XZZNTLNFQVAKFD-UHFFFAOYSA-N | $8.8{\times}10^{-4}$ $2.8{\times}10^{-4}$ $7.9{\times}10^{-4}$ | | Yaws (2003) Gharagheizi et al. (2012) Gharagheizi et al. (2010) | X Q Q | 237 246 |
| 1-methyl-3,5-diethylbenzene $C_{11}H_{16}$ [2050-24-0] HILAULICMJUOLK-UHFFFAOYSA-N | $9.5{\times}10^{-4}$ $8.7{\times}10^{-4}$ $5.8{\times}10^{-4}$ $2.0{\times}10^{-3}$ $2.2{\times}10^{-4}$ $7.9{\times}10^{-4}$ | | Yaws (2003) Wang et al. (2017) Wang et al. (2017) Wang et al. (2017) Gharagheizi et al. (2012) Gharagheizi et al. (2010) | X Q Q Q Q Q | 237 80, 238 80, 239 80, 240 246 |
| 1-methyl-2-butylbenzene $C_{11}H_{16}$ [1595-11-5] NUJILYKLNKQOOX-UHFFFAOYSA-N | $9.0{\times}10^{-4}$ $4.1{\times}10^{-4}$ $8.8{\times}10^{-4}$ | | Yaws (2003) Gharagheizi et al. (2012) Gharagheizi et al. (2010) | X Q Q | 237 246 |
| 1-methyl-3-butylbenzene $C_{11}H_{16}$ [1595-04-6] OAPCPUDMDJIBOQ-UHFFFAOYSA-N | $9.2{\times}10^{-4}$ $3.7{\times}10^{-4}$ $8.8{\times}10^{-4}$ | | Yaws (2003) Gharagheizi et al. (2012) Gharagheizi et al. (2010) | X Q Q | 237 246 |
| 1-methyl-4-butylbenzene $C_{11}H_{16}$ [1595-05-7] SBBKUBSYOVDBBC-UHFFFAOYSA-N | $9.1{\times}10^{-4}$ $4.0{\times}10^{-4}$ $8.8{\times}10^{-4}$ | | Yaws (2003) Gharagheizi et al. (2012) Gharagheizi et al. (2010) | X Q Q | 237 246 |
| 1-methyl-2-isobutylbenzene $C_{11}H_{16}$ [36301-29-8] XNMPJDZAHSMAMN-UHFFFAOYSA-N | $9.1{\times}10^{-4}$ $2.9{\times}10^{-4}$ $9.1{\times}10^{-4}$ | | Yaws (2003) Gharagheizi et al. (2012) Gharagheizi et al. (2010) | X Q Q | 237 246 |
| 1-methyl-3-isobutylbenzene $C_{11}H_{16}$ [5160-99-6] SDHYGAUOCHFYSR-UHFFFAOYSA-N | $9.3{\times}10^{-4}$ $2.8{\times}10^{-4}$ $9.1{\times}10^{-4}$ | | Yaws (2003) Gharagheizi et al. (2012) Gharagheizi et al. (2010) | X Q Q | 237 246 |
| 1-methyl-4-isobutylbenzene $C_{11}H_{16}$ [5161-04-6] VCGBZXLLPCGFQM-UHFFFAOYSA-N | $9.1{\times}10^{-4}$ $2.9{\times}10^{-4}$ $9.1{\times}10^{-4}$ | | Yaws (2003) Gharagheizi et al. (2012) Gharagheizi et al. (2010) | X Q Q | 237 246 |
| 1-methyl-2-*sec*-butylbenzene $C_{11}H_{16}$ [1595-06-8] AMBAWAHKHZZAAY-UHFFFAOYSA-N | $9.1{\times}10^{-4}$ $2.9{\times}10^{-4}$ $9.1{\times}10^{-4}$ | | Yaws (2003) Gharagheizi et al. (2012) Gharagheizi et al. (2010) | X Q Q | 237 246 |



Table A2.5: Mononuclear aromatics (...continued)

| Substance Formula (Trivial Name) [CAS Registry Number] InChIKey | $H_s^{cp}$ (at $T^\ominus$) $\left[\dfrac{\mathrm{mol}}{\mathrm{m^3\,Pa}}\right]$ | $\dfrac{\mathrm{d}\ln H_s^{cp}}{\mathrm{d}(1/T)}$ [K] | Reference | Type | Note |
|---|---|---|---|---|---|
| 1-methyl-3-*sec*-butylbenzene $C_{11}H_{16}$ [1772-10-7] RMNILBOMCXQZFC-UHFFFAOYSA-N | $9.3\times10^{-4}$ $2.8\times10^{-4}$ $9.1\times10^{-4}$ | | Yaws (2003) Gharagheizi et al. (2012) Gharagheizi et al. (2010) | X Q Q | 237 246 |
| 1-methyl-4-*sec*-butylbenzene $C_{11}H_{16}$ [1595-16-0] LWCFXYMSEGQWNB-UHFFFAOYSA-N | $9.0\times10^{-4}$ $3.0\times10^{-4}$ $9.1\times10^{-4}$ | | Yaws (2003) Gharagheizi et al. (2012) Gharagheizi et al. (2010) | X Q Q | 237 246 |
| 1-methyl-2-*tert*-butyl benzene $C_{11}H_{16}$ [1074-92-6] AXHVNJGQOJFMHT-UHFFFAOYSA-N | $7.2\times10^{-4}$ $3.4\times10^{-4}$ $8.2\times10^{-4}$ | | Yaws (2003) Gharagheizi et al. (2012) Gharagheizi et al. (2010) | X Q Q | 237 246 |
| 1-methyl-3-*tert*-butylbenzene $C_{11}H_{16}$ [1075-38-3] JTIAYWZZZOZUTK-UHFFFAOYSA-N | $8.2\times10^{-4}$ $2.5\times10^{-4}$ $8.2\times10^{-4}$ | | Yaws (2003) Gharagheizi et al. (2012) Gharagheizi et al. (2010) | X Q Q | 237 246 |
| 1-methyl-3,4-diethylbenzene $C_{11}H_{16}$ [13732-80-4] MJQJAQGFUBIGIK-UHFFFAOYSA-N | $9.2\times10^{-4}$ $2.4\times10^{-4}$ $7.9\times10^{-4}$ | | Yaws (2003) Gharagheizi et al. (2012) Gharagheizi et al. (2010) | X Q Q | 237 246 |
| 1-phenyl-2,2-dimethylpropane $C_{11}H_{16}$ [1007-26-7] CJGXJKVMUHXVHL-UHFFFAOYSA-N | $3.8\times10^{-4}$ $3.4\times10^{-4}$ $4.5\times10^{-4}$ | | Yaws (2003) Gharagheizi et al. (2012) Gharagheizi et al. (2010) | X Q Q | 237 246 |
| 1-phenyl-2-methylbutane $C_{11}H_{16}$ [3968-85-2] IFDLFCDWOFLKEB-UHFFFAOYSA-N | $4.0\times10^{-4}$ $4.5\times10^{-4}$ $5.5\times10^{-4}$ | | Yaws (2003) Gharagheizi et al. (2012) Gharagheizi et al. (2010) | X Q Q | 237 246 |
| 1-phenyl-3-methylbutane $C_{11}H_{16}$ [2049-94-7] XNXIYYFOYIUJIW-UHFFFAOYSA-N | $3.9\times10^{-4}$ $4.8\times10^{-4}$ $5.5\times10^{-4}$ | | Yaws (2003) Gharagheizi et al. (2012) Gharagheizi et al. (2010) | X Q Q | 237 246 |
| 2-phenyl-3-methylbutane $C_{11}H_{16}$ [4481-30-5] NQRMTOKLHZNAQH-UHFFFAOYSA-N | $4.1\times10^{-4}$ $3.6\times10^{-4}$ $5.3\times10^{-4}$ | | Yaws (2003) Gharagheizi et al. (2012) Gharagheizi et al. (2010) | X Q Q | 237 246 |
| 2-phenylpentane $C_{11}H_{16}$ [2719-52-0] LTHAIAJHDPJXLG-UHFFFAOYSA-N | $4.3\times10^{-4}$ $3.7\times10^{-4}$ $5.5\times10^{-4}$ | | Yaws (2003) Gharagheizi et al. (2012) Gharagheizi et al. (2010) | X Q Q | 237 246 |



Table A2.5: Mononuclear aromatics (...continued)

| Substance Formula (Trivial Name) [CAS Registry Number] InChIKey | $H_s^{cp}$ (at $T^\ominus$) $\left[\dfrac{\mathrm{mol}}{\mathrm{m^3\,Pa}}\right]$ | $\dfrac{\mathrm{d}\ln H_s^{cp}}{\mathrm{d}(1/T)}$ [K] | Reference | Type | Note |
|---|---|---|---|---|---|
| 3-phenylpentane | $4.3\times10^{-4}$ | | Yaws (2003) | X | 237 |
| $C_{11}H_{16}$ | $3.8\times10^{-4}$ | | Gharagheizi et al. (2012) | Q | |
| [1196-58-3] | $5.5\times10^{-4}$ | | Gharagheizi et al. (2010) | Q | 246 |
| PBWHJRFXUPLZDS-UHFFFAOYSA-N | | | | | |
| (1,1-dimethylpropyl)-benzene | $5.4\times10^{-4}$ | | Hine and Mookerjee (1975) | V | |
| $C_6H_5C_5H_{11}$ | $3.5\times10^{-4}$ | | Yaws (2003) | X | 237 |
| (*tert*-amylbenzene) | $4.1\times10^{-4}$ | | Gharagheizi et al. (2012) | Q | |
| [2049-95-8] | $4.9\times10^{-4}$ | | Raventos-Duran et al. (2010) | Q | 242, 243 |
| QHTJSSMHBLGUHV-UHFFFAOYSA-N | $6.2\times10^{-4}$ | | Raventos-Duran et al. (2010) | Q | 244 |
| | $4.9\times10^{-4}$ | | Raventos-Duran et al. (2010) | Q | 245 |
| | $4.5\times10^{-4}$ | | Gharagheizi et al. (2010) | Q | 246 |
| | $9.9\times10^{-4}$ | | Hilal et al. (2008) | Q | |
| | $5.8\times10^{-4}$ | | English and Carroll (2001) | Q | 230, 231 |
| | $4.2\times10^{-4}$ | | Suzuki et al. (1992) | Q | 232 |
| | $5.1\times10^{-4}$ | | Nirmalakhandan and Speece (1988) | Q | |
| hexylbenzene | $4.0\times10^{-4}$ | 6500 | Brockbank (2013) | L | 1 |
| $C_6H_5C_6H_{13}$ | $4.0\times10^{-4}$ | 6300 | Plyasunov and Shock (2000) | L | |
| [1077-16-3] | $3.5\times10^{-4}$ | | Duchowicz et al. (2020) | V | 186 |
| LTEQMZWBSYACLV-UHFFFAOYSA-N | $4.6\times10^{-4}$ | | Mackay et al. (2006a) | V | |
| | $4.6\times10^{-4}$ | | Mackay et al. (1992a) | V | |
| | $4.5\times10^{-4}$ | | Meylan and Howard (1991) | V | |
| | $5.1\times10^{-4}$ | | Eastcott et al. (1988) | V | |
| | $4.5\times10^{-4}$ | | Abraham (1984) | V | |
| | $7.7\times10^{-3}$ | 9000 | Ben-Naim and Wilf (1980) | V | 1 |
| | $4.1\times10^{-4}$ | 6300 | Owens et al. (1986) | T | |
| | $3.9\times10^{-4}$ | | Yaws (2003) | X | 237 |
| | $3.1\times10^{-3}$ | | Duchowicz et al. (2020) | Q | |
| | $6.2\times10^{-4}$ | | Gharagheizi et al. (2012) | Q | |
| | $5.3\times10^{-4}$ | | Gharagheizi et al. (2010) | Q | 246 |
| | $4.8\times10^{-4}$ | | Hilal et al. (2008) | Q | |
| | $4.3\times10^{-4}$ | | Modarresi et al. (2007) | Q | 67 |
| | $4.6\times10^{-4}$ | | Yaffe et al. (2003) | Q | 248, 249 |
| | $3.7\times10^{-4}$ | | Yao et al. (2002) | Q | 229 |
| | $4.2\times10^{-4}$ | | English and Carroll (2001) | Q | 230, 231 |
| | $5.0\times10^{-4}$ | | Nirmalakhandan et al. (1997) | Q | |
| | $4.0\times10^{-4}$ | | Meylan and Howard (1991) | Q | |
| | $3.9\times10^{-4}$ | | Yaws (1999) | ? | 21 |
| | $4.6\times10^{-4}$ | | Yaws and Yang (1992) | ? | 21 |
| | $4.3\times10^{-4}$ | | Abraham et al. (1990) | ? | |
| hexamethylbenzene | $8.6\times10^{-3}$ | | Hilal et al. (2008) | Q | |
| $C_{12}H_{18}$ | $1.5\times10^{-3}$ | | Modarresi et al. (2007) | Q | 67 |
| [87-85-4] | | | | | |
| YUWFEBAXEOLKSG-UHFFFAOYSA-N | | | | | |



Table A2.5: Mononuclear aromatics (...continued)

| Substance Formula (Trivial Name) [CAS Registry Number] InChIKey | $H_s^{cp}$ (at $T^{\ominus}$) $\left[\dfrac{\text{mol}}{\text{m}^3\,\text{Pa}}\right]$ | $\dfrac{\text{d}\ln H_s^{cp}}{\text{d}(1/T)}$ [K] | Reference | Type | Note |
|---|---|---|---|---|---|
| 1,2,3-triethylbenzene | $3.5\times10^{-4}$ | | Yaws (2003) | X | 237 |
| $C_{12}H_{18}$ | $1.4\times10^{-4}$ | | Gharagheizi et al. (2012) | Q | |
| [42205-08-3] | $6.0\times10^{-4}$ | | Gharagheizi et al. (2010) | Q | 246 |
| VIDOPANCAUPXNH-UHFFFAOYSA-N | | | | | |
| 1,2,4-triethylbenzene | $7.2\times10^{-4}$ | | Yaws (2003) | X | 237 |
| $C_{12}H_{18}$ | $1.4\times10^{-4}$ | | Gharagheizi et al. (2012) | Q | |
| [877-44-1] | $6.0\times10^{-4}$ | | Gharagheizi et al. (2010) | Q | 246 |
| WNLWIOJSURYFIB-UHFFFAOYSA-N | $7.2\times10^{-4}$ | | Yaws (1999) | ? | 21 |
| 1,3,5-triethylbenzene | $1.0\times10^{-3}$ | | Plyasunov and Shock (2000) | L | |
| $C_{12}H_{18}$ | $3.4\times10^{-4}$ | | Yaws (2003) | X | 237 |
| [102-25-0] | $1.4\times10^{-4}$ | | Gharagheizi et al. (2012) | Q | |
| WJYMPXJVHNDZHD-UHFFFAOYSA-N | $6.0\times10^{-4}$ | | Gharagheizi et al. (2010) | Q | 246 |
| 4-*tert*-butyl-*o*-xylene | $5.8\times10^{-4}$ | | Zhang et al. (2010) | Q | 287, 288 |
| $C_{12}H_{18}$ | $7.2\times10^{-4}$ | | Zhang et al. (2010) | Q | 287, 289 |
| [7397-06-0] | $9.0\times10^{-4}$ | | Zhang et al. (2010) | Q | 287, 290 |
| QRPPSTNABSMSCS-UHFFFAOYSA-N | $2.7\times10^{-4}$ | | Zhang et al. (2010) | Q | 287, 291 |
| 1-(1,1-dimethylethyl)-3,5-dimethylbenzene | $5.8\times10^{-4}$ | | Zhang et al. (2010) | Q | 287, 288 |
| $C_{12}H_{18}$ | $4.5\times10^{-4}$ | | Zhang et al. (2010) | Q | 287, 289 |
| [98-19-1] | $7.7\times10^{-4}$ | | Zhang et al. (2010) | Q | 287, 290 |
| FZSPYHREEHYLCB-UHFFFAOYSA-N | $2.7\times10^{-4}$ | | Zhang et al. (2010) | Q | 287, 291 |
| *o*-diisopropylbenzene | $4.8\times10^{-4}$ | | HSDB (2015) | Q | 99 |
| $C_{12}H_{18}$ | | | | | |
| [25321-09-9] | | | | | |
| OKIRBHVFJGXOIS-UHFFFAOYSA-N | | | | | |
| *m*-diisopropylbenzene | $9.5\times10^{-4}$ | | Yaws (2003) | X | 237 |
| $C_{12}H_{18}$ | $7.2\times10^{-4}$ | | Gharagheizi et al. (2010) | Q | 246 |
| [99-62-7] | $9.6\times10^{-4}$ | | Yaws (1999) | ? | 21 |
| UNEATYXSUBPPKP-UHFFFAOYSA-N | | | | | |
| *p*-diisopropylbenzene | $1.0\times10^{-3}$ | | Yaws (2003) | X | 237 |
| $C_{12}H_{18}$ | $1.8\times10^{-4}$ | | Gharagheizi et al. (2012) | Q | |
| [100-18-5] | $7.2\times10^{-4}$ | | Gharagheizi et al. (2010) | Q | 246 |
| SPPWGCYEYAMHDT-UHFFFAOYSA-N | $1.0\times10^{-3}$ | | Yaws (1999) | ? | 21 |
| 4-phenylcyclohexene | $7.9\times10^{-3}$ | | Ebert et al. (2023) | ? | 318 |
| $C_{12}H_{14}$ | | | | | |
| [4994-16-5] | | | | | |
| XWCWNUSFQVJNDI-UHFFFAOYSA-N | | | | | |





Table A2.5: Mononuclear aromatics (...continued)

| Substance Formula (Trivial Name) [CAS Registry Number] InChIKey | $H_s^{cp}$ (at $T^{\ominus}$) $\left[\dfrac{\mathrm{mol}}{\mathrm{m^3\,Pa}}\right]$ | $\dfrac{\mathrm{d}\ln H_s^{cp}}{\mathrm{d}(1/T)}$ [K] | Reference | Type | Note |
|---|---|---|---|---|---|
| heptylbenzene $C_6H_5C_7H_{15}$ [1078-71-3] LBNXAWYDQUGHGX-UHFFFAOYSA-N | $2.7\times10^{-4}$ $6.5\times10^{-4}$ $2.2\times10^{-2}$ $6.5\times10^{-4}$ $3.1\times10^{-3}$ $1.0\times10^{-3}$ $5.0\times10^{-4}$ $3.9\times10^{-4}$ $3.5\times10^{-4}$ $1.5\times10^{-3}$ | 11000 | Brockbank (2013) Duchowicz et al. (2020) Ben-Naim and Wilf (1980) Yaws (2003) Duchowicz et al. (2020) Gharagheizi et al. (2012) Gharagheizi et al. (2010) Hilal et al. (2008) Yao et al. (2002) Yaws (1999) | L V V X Q Q Q Q Q ? | 186 1 237 246 229 21 |
| 5-*tert*-butyl-1,2,3-trimethylbenzene $C_{13}H_{20}$ [98-23-7] ZQVJKYPEIPJEIP-UHFFFAOYSA-N | $5.3\times10^{-4}$ $9.2\times10^{-4}$ $9.0\times10^{-4}$ $1.5\times10^{-4}$ | | Zhang et al. (2010) Zhang et al. (2010) Zhang et al. (2010) Zhang et al. (2010) | Q Q Q Q | 287, 288 287, 289 287, 290 287, 291 |
| octylbenzene $C_6H_5C_8H_{17}$ [2189-60-8] CDKDZKXSXLNROY-UHFFFAOYSA-N | $1.8\times10^{-4}$ $2.3\times10^{-4}$ $5.4\times10^{-2}$ $8.7\times10^{-4}$ $3.1\times10^{-3}$ $1.7\times10^{-3}$ $4.9\times10^{-4}$ $3.2\times10^{-4}$ $2.0\times10^{-3}$ | 12000 | Brockbank (2013) Duchowicz et al. (2020) Ben-Naim and Wilf (1980) Yaws (2003) Duchowicz et al. (2020) Gharagheizi et al. (2012) Gharagheizi et al. (2010) Hilal et al. (2008) Yaws (1999) | L V V X Q Q Q Q ? | 186 1 237 246 21 |
| nonylbenzene $C_{15}H_{24}$ [1081-77-2] LIXVMPBOGDCSRM-UHFFFAOYSA-N | $1.4\times10^{-4}$ $1.8\times10^{-3}$ $3.3\times10^{-3}$ | | Brockbank (2013) Gharagheizi et al. (2012) Yaws (1999) | L Q ? | 21 |
| 3,5-di-*tert*-butyltoluene $C_{15}H_{24}$ [15181-11-0] WIXDSJRJFDWTNY-UHFFFAOYSA-N | $3.7\times10^{-3}$ | 9100 | Hiatt (2013) | M | |
| 1,3,5-tris(1-methylethyl)benzene $C_{15}H_{24}$ [717-74-8] VUMCUSHVMYIRMB-UHFFFAOYSA-N | $2.5\times10^{-4}$ $1.8\times10^{-4}$ $5.2\times10^{-4}$ $2.6\times10^{-4}$ | | Zhang et al. (2010) Zhang et al. (2010) Zhang et al. (2010) Zhang et al. (2010) | Q Q Q Q | 287, 288 287, 289 287, 290 287, 291 |
| ethyl(phenylethyl)-benzene $C_{16}H_{18}$ [64800-83-5] BDEIYMXBPHSOSG-UHFFFAOYSA-N | $1.1\times10^{-2}$ $1.2\times10^{-2}$ $6.4\times10^{-2}$ $1.8\times10^{-2}$ | | Zhang et al. (2010) Zhang et al. (2010) Zhang et al. (2010) Zhang et al. (2010) | Q Q Q Q | 287, 288 287, 289 287, 290 287, 291 |
| 4-(1-phenylethyl)-*m*-xylene $C_{16}H_{18}$ [6165-52-2] JOUBGGHXBLOLFY-UHFFFAOYSA-N | $1.3\times10^{-2}$ $1.6\times10^{-2}$ $5.2\times10^{-2}$ $1.5\times10^{-2}$ | | Zhang et al. (2010) Zhang et al. (2010) Zhang et al. (2010) Zhang et al. (2010) | Q Q Q Q | 287, 288 287, 289 287, 290 287, 291 |



Table A2.5: Mononuclear aromatics (...continued)

| Substance Formula (Trivial Name) [CAS Registry Number] InChIKey | $H_s^{cp}$ (at $T^{\ominus}$) $\left[\dfrac{\text{mol}}{\text{m}^3\,\text{Pa}}\right]$ | $\dfrac{\mathrm{d}\ln H_s^{cp}}{\mathrm{d}(1/T)}$ [K] | Reference | Type | Note |
|---|---|---|---|---|---|
| decylbenzene $C_{16}H_{26}$ (1-phenyldecane) [104-72-3] UZILCZKGXMQEQR-UHFFFAOYSA-N | $7.6\times10^{-5}$ $8.6\times10^{-5}$ $6.5\times10^{-5}$ $1.3\times10^{-4}$ $8.6\times10^{-5}$ $3.1\times10^{-3}$ $1.3\times10^{-4}$ $1.4\times10^{-4}$ $3.4\times10^{-4}$ $2.8\times10^{-4}$ $7.9\times10^{-3}$ | | Brockbank (2013) Plyasunov and Shock (2000) Duchowicz et al. (2020) HSDB (2015) Sherblom et al. (1992) Duchowicz et al. (2020) Zhang et al. (2010) Zhang et al. (2010) Zhang et al. (2010) Zhang et al. (2010) Yaws (1999) | L L V V V Q Q Q Q Q ? | 186 287, 288 287, 289 287, 290 287, 291 21 |
| 2-phenyldecane $C_{16}H_{26}$ [4537-13-7] DDTJIQUCOLHYDL-UHFFFAOYSA-N | $1.0\times10^{-4}$ | | Sherblom et al. (1992) | V | |
| 3-phenyldecane $C_{16}H_{26}$ [4621-36-7] PYVIFMPVFLOTLN-UHFFFAOYSA-N | $1.1\times10^{-4}$ | | Sherblom et al. (1992) | V | |
| 4-phenyldecane $C_{16}H_{26}$ [4537-12-6] QTDBKYLPIZUTFN-UHFFFAOYSA-N | $9.2\times10^{-5}$ | | Sherblom et al. (1992) | V | |
| 5-phenyldecane $C_{16}H_{26}$ [4537-11-5] CDOBABYRHNVZQG-UHFFFAOYSA-N | $8.4\times10^{-5}$ | | Sherblom et al. (1992) | V | |
| undecylbenzene $C_{17}H_{28}$ (1-phenylundecane) [6742-54-7] XBEADGFTLHRJRB-UHFFFAOYSA-N | $9.9\times10^{-5}$ | | HSDB (2015) | Q | 99 |
| 2-phenylundecane $C_{17}H_{28}$ [4536-88-3] YHJBCRBYDBNEIK-UHFFFAOYSA-N | $8.2\times10^{-5}$ | | Sherblom et al. (1992) | V | |
| 3-phenylundecane $C_{17}H_{28}$ [4536-87-2] NVHBFHMWJJMQTG-UHFFFAOYSA-N | $9.0\times10^{-5}$ | | Sherblom et al. (1992) | V | |





Table A2.5: Mononuclear aromatics (...continued)

| Substance Formula (Trivial Name) [CAS Registry Number] InChIKey | $H_s^{cp}$ (at $T^{\ominus}$) $\left[\dfrac{\text{mol}}{\text{m}^3\,\text{Pa}}\right]$ | $\dfrac{\text{d}\ln H_s^{cp}}{\text{d}(1/T)}$ [K] | Reference | Type | Note |
|---|---|---|---|---|---|
| 4-phenylundecane C$_{17}$H$_{28}$ [4536-86-1] NSQAXMRLBNXEHK-UHFFFAOYSA-N | $5.9\times10^{-5}$ | | Sherblom et al. (1992) | V | |
| 5-phenylundecane C$_{17}$H$_{28}$ [4537-15-9] RRPCXIBGXYGQNC-UHFFFAOYSA-N | $6.1\times10^{-5}$ | | Sherblom et al. (1992) | V | |
| 6-phenylundecane C$_{17}$H$_{28}$ [4537-14-8] WCABIRIFXVXGQH-UHFFFAOYSA-N | $6.4\times10^{-5}$ | | Sherblom et al. (1992) | V | |
| dodecylbenzene C$_{18}$H$_{30}$ (1-phenyldodecane) [123-01-3] KWKXNDCHNDYVRT-UHFFFAOYSA-N | $7.6\times10^{-5}$ | | HSDB (2015) | Q | 99 |
| 2-phenyldodecane C$_{18}$H$_{30}$ [2719-61-1] VRPRIAVYSREHAN-UHFFFAOYSA-N | $1.1\times10^{-4}$ | | Sherblom et al. (1992) | V | |
| 3-phenyldodecane C$_{18}$H$_{30}$ [2400-00-2] PGVOXXHNGYYHHB-UHFFFAOYSA-N | $1.3\times10^{-4}$ | | Sherblom et al. (1992) | V | |
| 4-phenyldodecane C$_{18}$H$_{30}$ [2719-64-4] RHDHXBLZBVAPTL-UHFFFAOYSA-N | $8.3\times10^{-5}$ | | Sherblom et al. (1992) | V | |
| 5-phenyldodecane C$_{18}$H$_{30}$ [2719-63-3] NPAWGLOPXKCTCV-UHFFFAOYSA-N | $7.8\times10^{-5}$ | | Sherblom et al. (1992) | V | |
| 6-phenyldodecane C$_{18}$H$_{30}$ [2719-62-2] ZYHJQFMTTFCBKH-UHFFFAOYSA-N | $6.0\times10^{-5}$ | | Sherblom et al. (1992) | V | |
| tridecylbenzene C$_{19}$H$_{32}$ (1-phenyltridecane) [123-02-4] MCVUKOYZUCWLQQ-UHFFFAOYSA-N | $1.1\times10^{-4}$ $5.5\times10^{-5}$ | | Sherblom et al. (1992) HSDB (2015) | V Q | 99 |



Table A2.5: Mononuclear aromatics (...continued)

| Substance Formula (Trivial Name) [CAS Registry Number] InChIKey | $H_s^{cp}$ (at $T^{\ominus}$) $\left[\dfrac{\text{mol}}{\text{m}^3\,\text{Pa}}\right]$ | $\dfrac{\text{d}\ln H_s^{cp}}{\text{d}(1/T)}$ [K] | Reference | Type | Note |
|---|---|---|---|---|---|
| 2-phenyltridecane $C_{19}H_{32}$ [4534-53-6] FCXPVFLEDIQLLO-UHFFFAOYSA-N | $2.7\times10^{-4}$ | | Sherblom et al. (1992) | V | |
| 3-phenyltridecane $C_{19}H_{32}$ [4534-52-5] VZZMNLVGDGMQQV-UHFFFAOYSA-N | $2.1\times10^{-4}$ | | Sherblom et al. (1992) | V | |
| 4-phenyltridecane $C_{19}H_{32}$ [4534-51-4] RZGVZPAWCGDMCK-UHFFFAOYSA-N | $2.0\times10^{-4}$ | | Sherblom et al. (1992) | V | |
| 5-phenyltridecane $C_{19}H_{32}$ [4534-50-3] MZTIRLOLMGVVEK-UHFFFAOYSA-N | $1.5\times10^{-4}$ | | Sherblom et al. (1992) | V | |
| 6-phenyltridecane $C_{19}H_{32}$ [4534-49-0] OTSYFFDVDLHIKX-UHFFFAOYSA-N | $1.5\times10^{-4}$ | | Sherblom et al. (1992) | V | |
| tetradecylbenzene $C_{20}H_{34}$ (1-phenyltetradecane) [1459-10-5] JZALLXAUNPOCEU-UHFFFAOYSA-N | $4.2\times10^{-5}$ | | HSDB (2015) | Q | 99 |
| 2-phenyltetradecane $C_{20}H_{34}$ [4534-59-2] GDFUGKICRHMMOT-UHFFFAOYSA-N | $6.7\times10^{-4}$ | | Sherblom et al. (1992) | V | |
| 3-phenyltetradecane $C_{20}H_{34}$ [4534-58-1] ILOABSZOHZMWLD-UHFFFAOYSA-N | $6.2\times10^{-4}$ | | Sherblom et al. (1992) | V | |
| 4-phenyltetradecane $C_{20}H_{34}$ [4534-57-0] YXLRHYCRTUAPMC-UHFFFAOYSA-N | $4.0\times10^{-4}$ | | Sherblom et al. (1992) | V | |
| 5-phenyltetradecane $C_{20}H_{34}$ [4534-56-9] RAWFVRXBPRHWLJ-UHFFFAOYSA-N | $5.0\times10^{-4}$ | | Sherblom et al. (1992) | V | |



Table A2.5: Mononuclear aromatics (...continued)

| Substance<br>Formula<br>(Trivial Name)<br>[CAS Registry Number]<br>InChIKey | $H_s^{cp}$<br>(at $T^\ominus$)<br>$\left[\dfrac{\text{mol}}{\text{m}^3\,\text{Pa}}\right]$ | $\dfrac{\mathrm{d}\ln H_s^{cp}}{\mathrm{d}(1/T)}$<br><br>[K] | Reference | Type | Note |
|---|---|---|---|---|---|
| 6-phenyltetradecane<br>C$_{20}$H$_{34}$<br>[4534-55-8]<br>MTDIFFBFSKQPIA-UHFFFAOYSA-N | $3.6\times10^{-4}$ | | Sherblom et al. (1992) | V | |
| pentadecylbenzene<br>C$_{21}$H$_{36}$<br>[2131-18-2]<br>JIRNEODMTPGRGV-UHFFFAOYSA-N | $1.2\times10^{-5}$ | | HSDB (2015) | Q | 99 |
| ethenylbenzene<br>C$_8$H$_8$<br>(styrene)<br>[100-42-5]<br>PPBRXRYQALVLMV-UHFFFAOYSA-N | $3.6\times10^{-3}$ | 3700 | Brockbank (2013) | L | 1 |
| | $2.9\times10^{-3}$ | 4200 | Plyasunov and Shock (2000) | L | |
| | $2.7\times10^{-3}$ | | Kim and Kim (2014) | M | |
| | $4.4\times10^{-3}$ | 4600 | Hiatt (2013) | M | |
| | $3.4\times10^{-3}$ | | Dohnal and Hovorka (1999) | M | |
| | $2.5\times10^{-3}$ | | Welke et al. (1998) | M | |
| | $3.8\times10^{-3}$ | 4100 | Kondoh and Nakajima (1997) | M | |
| | $2.9\times10^{-3}$ | 4800 | Bissonette et al. (1990) | M | |
| | $1.8\times10^{-3}$ | | Sato and Nakajima (1979a) | M | 14 |
| | $3.6\times10^{-3}$ | | Lide and Frederikse (1995) | V | |
| | $3.3\times10^{-3}$ | | Abraham et al. (1994a) | V | |
| | $3.3\times10^{-3}$ | | Mackay et al. (1993) | V | |
| | $3.8\times10^{-3}$ | | Yaws (2003) | X | 258 |
| | $3.8\times10^{-3}$ | | Yaws (2003) | X | 237 |
| | $3.8\times10^{-3}$ | 4200 | Goldstein (1982) | X | 298 |
| | $3.8\times10^{-3}$ | 3800 | Fogg and Sangster (2003) | C | |
| | $3.3\times10^{-3}$ | | Dupeux et al. (2022) | Q | 259 |
| | $1.3\times10^{-3}$ | | Keshavarz et al. (2022) | Q | |
| | $1.1\times10^{-2}$ | | Duchowicz et al. (2020) | Q | 299 |
| | $8.7\times10^{-3}$ | | Wang et al. (2017) | Q | 80, 238 |
| | $2.6\times10^{-3}$ | | Wang et al. (2017) | Q | 80, 239 |
| | $7.6\times10^{-3}$ | | Wang et al. (2017) | Q | 80, 240 |
| | $1.6\times10^{-2}$ | | Gharagheizi et al. (2012) | Q | |
| | $4.9\times10^{-3}$ | | Raventos-Duran et al. (2010) | Q | 242, 243 |
| | $2.5\times10^{-3}$ | | Raventos-Duran et al. (2010) | Q | 244 |
| | $3.9\times10^{-3}$ | | Raventos-Duran et al. (2010) | Q | 245 |
| | $2.6\times10^{-3}$ | | Gharagheizi et al. (2010) | Q | 246 |
| | $3.2\times10^{-3}$ | | Hilal et al. (2008) | Q | |
| | $4.4\times10^{-3}$ | | Modarresi et al. (2007) | Q | 67 |
| | | 4800 | Kühne et al. (2005) | Q | |
| | $3.8\times10^{-3}$ | | Yaffe et al. (2003) | Q | 248, 249 |
| | $3.3\times10^{-3}$ | | English and Carroll (2001) | Q | 230, 231 |
| | $1.2\times10^{-4}$ | | Katritzky et al. (1998) | Q | |
| | $3.7\times10^{-3}$ | | Nirmalakhandan et al. (1997) | Q | |
| | $3.6\times10^{-3}$ | | Duchowicz et al. (2020) | ? | 185, 21 |
| | | 3700 | Kühne et al. (2005) | ? | |
| | $3.8\times10^{-3}$ | | Yaws (1999) | ? | 21 |
| | $1.9\times10^{-3}$ | | Abraham and Weathersby (1994) | ? | 21 |





Table A2.5: Mononuclear aromatics (... continued)

| Substance Formula (Trivial Name) [CAS Registry Number] InChIKey | $H_s^{cp}$ (at $T^\ominus$) $\left[\dfrac{\text{mol}}{\text{m}^3\,\text{Pa}}\right]$ | $\dfrac{\mathrm{d}\ln H_s^{cp}}{\mathrm{d}(1/T)}$ [K] | Reference | Type | Note |
|---|---|---|---|---|---|
| | $3.7\times10^{-3}$ | | Yaws and Yang (1992) | ? | 21 |
| | | | Shiu and Ma (2000) | W | 360 |
| (E)-1-propenylbenzene $C_9H_{10}$ [873-66-5] QROGIFZRVHSFLM-QHHAFSJGSA-N | $2.9\times10^{-3}$ | | Hilal et al. (2008) | Q | |
| 1-propenylbenzene $C_9H_{10}$ [637-50-3] QROGIFZRVHSFLM-UHFFFAOYSA-N | $3.7\times10^{-3}$ | | HSDB (2015) | Q | 99 |
| 2-propenylbenzene $C_9H_{10}$ (allylbenzene) [300-57-2] HJWLCRVIBGQPNF-UHFFFAOYSA-N | $1.4\times10^{-3}$ $2.2\times10^{-3}$ $1.5\times10^{-3}$ $2.9\times10^{-3}$ $1.4\times10^{-3}$ | | Sato and Nakajima (1979a) Hilal et al. (2008) Yaffe et al. (2003) Nirmalakhandan et al. (1997) Abraham and Weathersby (1994) | M Q Q Q ? | 14 248, 249 21 |
| 1-ethenyl-3-methylbenzene $C_9H_{10}$ (m-methylstyrene) [100-80-1] JZHGRUMIRATHIU-UHFFFAOYSA-N | $3.0\times10^{-3}$ $7.0\times10^{-3}$ $3.8\times10^{-3}$ $3.1\times10^{-3}$ $1.6\times10^{-3}$ $3.0\times10^{-3}$ $2.6\times10^{-3}$ | | Yaws (2003) Gharagheizi et al. (2012) Gharagheizi et al. (2010) Hilal et al. (2008) Modarresi et al. (2007) Yaws (1999) Yaws and Yang (1992) | X Q Q Q Q ? ? | 237 246 67 21 21 |
| 1-ethenyl-4-methylbenzene $C_9H_{10}$ (p-methylstyrene) [622-97-9] JLBJTVDPSNHSKJ-UHFFFAOYSA-N | $3.1\times10^{-3}$ $3.1\times10^{-3}$ $3.1\times10^{-3}$ $5.5\times10^{-3}$ $7.6\times10^{-3}$ $3.1\times10^{-3}$ $2.5\times10^{-3}$ $3.1\times10^{-3}$ $3.8\times10^{-3}$ $3.4\times10^{-3}$ $2.8\times10^{-3}$ $1.5\times10^{-3}$ $3.1\times10^{-3}$ $3.5\times10^{-3}$ | | Duchowicz et al. (2020) HSDB (2015) Yaws (2003) Duchowicz et al. (2020) Gharagheizi et al. (2012) Raventos-Duran et al. (2010) Raventos-Duran et al. (2010) Raventos-Duran et al. (2010) Gharagheizi et al. (2010) Hilal et al. (2008) Modarresi et al. (2007) Yao et al. (2002) Yaws (1999) Yaws and Yang (1992) | V V X Q Q Q Q Q Q Q Q Q ? ? | 186 237 242, 243 244 245 246 67 229 21 21 |
| (1-methylethenyl)-benzene $C_9H_{10}$ ($\alpha$-methyl styrene) [98-83-9] XYLMUPLGERFSHI-UHFFFAOYSA-N | $3.9\times10^{-3}$ $3.8\times10^{-3}$ $3.3\times10^{-3}$ $3.6\times10^{-3}$ $2.4\times10^{-3}$ $3.9\times10^{-3}$ | | Duchowicz et al. (2020) HSDB (2015) Abraham et al. (1994a) Duchowicz et al. (2020) Hilal et al. (2008) English and Carroll (2001) | V V V Q Q Q | 186 230, 231 |



Table A2.5: Mononuclear aromatics (. . . continued)

| Substance<br>Formula<br>(Trivial Name)<br>[CAS Registry Number]<br>InChIKey | $H_s^{cp}$<br>(at $T^{\ominus}$)<br>$\left[\dfrac{\mathrm{mol}}{\mathrm{m^3\,Pa}}\right]$ | $\dfrac{\mathrm{d}\ln H_s^{cp}}{\mathrm{d}(1/T)}$<br>[K] | Reference | Type | Note |
|---|---|---|---|---|---|
| phenylacetylene | $1.6{\times}10^{-2}$ | | Duchowicz et al. (2020) | V | 186 |
| $C_8H_6$ | $5.7{\times}10^{-3}$ | | Yaws (2003) | X | 237 |
| [536-74-3] | $2.2{\times}10^{-2}$ | | Duchowicz et al. (2020) | Q | |
| UEXCJVNBTNXOEH-UHFFFAOYSA-N | $2.0{\times}10^{-2}$ | | Gharagheizi et al. (2012) | Q | |
| | $5.0{\times}10^{-3}$ | | Gharagheizi et al. (2010) | Q | 246 |
| | $3.9{\times}10^{-3}$ | | Hilal et al. (2008) | Q | |
| | $6.5{\times}10^{-3}$ | | Modarresi et al. (2007) | Q | 67 |
| | $1.5{\times}10^{-3}$ | | Yaffe et al. (2003) | Q | 248, 249 |
| $\alpha$-methylstyrene dimer | $1.1{\times}10^{-2}$ | | HSDB (2015) | Q | 99 |
| $C_{18}H_{20}$ | $5.7{\times}10^{-3}$ | | Zhang et al. (2010) | Q | 287, 288 |
| [6144-04-3] | $7.2{\times}10^{-3}$ | | Zhang et al. (2010) | Q | 287, 289 |
| ATSCZMXECPRPEA-BUHFOSPRSA-N | $2.4{\times}10^{-1}$ | | Zhang et al. (2010) | Q | 287, 290 |
| | $9.0{\times}10^{-2}$ | | Zhang et al. (2010) | Q | 287, 291 |





## A2.6 Terpenes and terpenoids

Table A2.6: Terpenes and terpenoids

| Substance<br>Formula<br>(Trivial Name)<br>[CAS Registry Number]<br><small>InChIKey</small> | $H_s^{cp}$<br>(at $T^{\ominus}$)<br>$\left[\dfrac{\text{mol}}{\text{m}^3\,\text{Pa}}\right]$ | $\dfrac{\text{d}\ln H_s^{cp}}{\text{d}(1/T)}$<br><br>[K] | Reference | Type | Note |
|---|---|---|---|---|---|
| 1-methyl-4-(1-methylethyl)-cyclohexane | $5.6\times10^{-6}$ | | Duchowicz et al. (2020) | V | 186 |
| $C_{10}H_{20}$ | $5.6\times10^{-6}$ | | Copolovici and Niinemets (2005) | V | |
| (*p*-menthane) | $2.1\times10^{-4}$ | | Duchowicz et al. (2020) | Q | |
| [99-82-1] | | | | | |
| <small>CFJYNSNXFXLKNS-UHFFFAOYSA-N</small> | | | | | |
| $\alpha$-pinene | $7.0\times10^{-5}$ | 4700 | Plyasunov and Shock (2000) | L | |
| $C_{10}H_{16}$ | $2.9\times10^{-4}$ | 1800 | Leng et al. (2013) | M | |
| [80-56-8] | $7.4\times10^{-5}$ | 4400 | Copolovici and Niinemets (2005) | M | |
| <small>GRWFGVWFFZKLTI-UHFFFAOYSA-N</small> | $5.8\times10^{-4}$ | | Karl et al. (2003) | M | 87 |
| | $1.4\times10^{-2}$ | | van Ruth et al. (2002) | M | 14 |
| | $7.4\times10^{-2}$ | | van Ruth and Villeneuve (2002) | M | 14, 361 |
| | $2.1\times10^{-3}$ | | van Ruth et al. (2001) | M | 14 |
| | $7.0\times10^{-5}$ | | Fichan et al. (1999) | M | |
| | $4.7\times10^{-5}$ | | Falk et al. (1990) | M | 14 |
| | $3.4\times10^{-5}$ | | Duchowicz et al. (2020) | V | 186 |
| | $3.4\times10^{-5}$ | | HSDB (2015) | V | |
| | $7.4\times10^{-5}$ | | Copolovici and Niinemets (2005) | V | |
| | $7.4\times10^{-5}$ | | Niinemets and Reichstein (2002) | V | |
| | $2.8\times10^{-5}$ | 10000 | Li et al. (1998) | V | |
| | $3.5\times10^{-5}$ | | Hilal et al. (2008) | C | |
| | $2.2\times10^{-4}$ | | Dupeux et al. (2022) | Q | 259 |
| | $3.0\times10^{-4}$ | | Duchowicz et al. (2020) | Q | |
| | $7.6\times10^{-4}$ | | Wang et al. (2017) | Q | 80, 238 |
| | $2.4\times10^{-5}$ | | Wang et al. (2017) | Q | 80, 239 |
| | $5.6\times10^{-4}$ | | Wang et al. (2017) | Q | 80, 240 |
| | $3.1\times10^{-5}$ | | Hilal et al. (2008) | Q | |
| $\beta$-pinene | $2.1\times10^{-4}$ | | Plyasunov and Shock (2000) | L | |
| $C_{10}H_{16}$ | $1.6\times10^{-4}$ | | Helburn et al. (2008) | M | |
| [127-91-3] | $1.5\times10^{-4}$ | 4500 | Copolovici and Niinemets (2005) | M | |
| <small>WTARULDDTDQWMU-UHFFFAOYSA-N</small> | $4.9\times10^{-4}$ | | Karl et al. (2003) | M | 87 |
| | $4.7\times10^{-5}$ | | Falk et al. (1990) | M | 14 |
| | $1.5\times10^{-4}$ | | Copolovici and Niinemets (2005) | V | |
| | $1.5\times10^{-4}$ | | Niinemets and Reichstein (2002) | V | |
| | $4.3\times10^{-4}$ | | Dupeux et al. (2022) | Q | 259 |
| | $4.5\times10^{-4}$ | | Wang et al. (2017) | Q | 80, 238 |
| | $3.2\times10^{-5}$ | | Wang et al. (2017) | Q | 80, 239 |
| | $1.1\times10^{-3}$ | | Wang et al. (2017) | Q | 80, 240 |
| | $6.2\times10^{-5}$ | | HSDB (2015) | Q | 99 |



Table A2.6: Terpenes and terpenoids (...continued)

| Substance Formula (Trivial Name) [CAS Registry Number] InChIKey | $H_s^{cp}$ (at $T^{\ominus}$) $\left[\dfrac{\mathrm{mol}}{\mathrm{m}^3\,\mathrm{Pa}}\right]$ | $\dfrac{\mathrm{d}\ln H_s^{cp}}{\mathrm{d}(1/T)}$ [K] | Reference | Type | Note |
|---|---|---|---|---|---|
| 1-methyl-4-(1-methylethyl)-1,3-cyclohexadiene | $5.1\times10^{-4}$ | | Plyasunov and Shock (2000) | L | |
| $C_{10}H_{16}$ | $3.8\times10^{-4}$ | | Schuhfried et al. (2015) | M | |
| ($\alpha$-terpinene) | $2.9\times10^{-4}$ | 4800 | Copolovici and Niinemets (2005) | M | |
| [99-86-5] | $4.5\times10^{-4}$ | | Karl et al. (2003) | M | 87 |
| YHQGMYUVUMAZJR-UHFFFAOYSA-N | $2.8\times10^{-4}$ | | Copolovici and Niinemets (2005) | V | |
| | $5.1\times10^{-4}$ | | Niinemets and Reichstein (2002) | V | |
| | $5.4\times10^{-4}$ | | Dupeux et al. (2022) | Q | 259 |
| 1-methyl-4-(1-methylethyl)-1,4-cyclohexadiene | $5.7\times10^{-4}$ | 4300 | Plyasunov and Shock (2000) | L | |
| $C_{10}H_{16}$ | $4.5\times10^{-4}$ | | Schuhfried et al. (2015) | M | |
| ($\gamma$-terpinene) | $3.8\times10^{-4}$ | 4800 | Copolovici and Niinemets (2005) | M | |
| [99-85-4] | $4.4\times10^{-4}$ | | Duchowicz et al. (2020) | V | 186 |
| YKFLAYDHMOASIY-UHFFFAOYSA-N | $3.8\times10^{-4}$ | | Copolovici and Niinemets (2005) | V | |
| | $2.8\times10^{-4}$ | | Niinemets and Reichstein (2002) | V | |
| | $5.4\times10^{-4}$ | 8000 | Li et al. (1998) | V | |
| | $6.2\times10^{-4}$ | | Dupeux et al. (2022) | Q | 259 |
| | $2.1\times10^{-4}$ | | Duchowicz et al. (2020) | Q | |
| | $3.4\times10^{-4}$ | | Katritzky et al. (1998) | Q | |
| 1-methyl-4-(1-methylethenyl)-cyclohexene | $4.7\times10^{-4}$ | 4700 | Plyasunov and Shock (2000) | L | |
| $C_{10}H_{16}$ | $4.8\times10^{-4}$ | 4600 | Leng et al. (2013) | M | |
| (limonene) | $2.6\times10^{-4}$ | | Copolovici and Niinemets (2007) | M | |
| [138-86-3] | $7.0\times10^{-4}$ | | Fichan et al. (1999) | M | |
| XMGQYMWWDOXHJM-UHFFFAOYSA-N | $2.4\times10^{-4}$ | | Welke et al. (1998) | M | |
| | $7.0\times10^{-4}$ | | Falk et al. (1990) | M | 14 |
| | $3.1\times10^{-4}$ | | Duchowicz et al. (2020) | V | 186 |
| | $3.1\times10^{-4}$ | | HSDB (2015) | V | |
| | $3.5\times10^{-4}$ | | Copolovici and Niinemets (2005) | V | |
| | $6.4\times10^{-4}$ | 3000 | van Roon et al. (2005) | V | |
| | $3.5\times10^{-4}$ | | Niinemets and Reichstein (2002) | V | |
| | $1.7\times10^{-4}$ | 10000 | Li et al. (1998) | V | |
| | $7.3\times10^{-4}$ | | Dupeux et al. (2022) | Q | 259 |
| | $6.4\times10^{-4}$ | | Duchowicz et al. (2020) | Q | |
| | $5.9\times10^{-4}$ | | Wang et al. (2017) | Q | 80, 238 |
| | $1.1\times10^{-4}$ | | Wang et al. (2017) | Q | 80, 239 |
| | $1.5\times10^{-3}$ | | Wang et al. (2017) | Q | 80, 240 |
| | $1.1\times10^{-4}$ | | Hilal et al. (2008) | Q | |
| | $1.9\times10^{-4}$ | | Modarresi et al. (2007) | Q | 67 |





Table A2.6: Terpenes and terpenoids (...continued)

| Substance<br>Formula<br>(Trivial Name)<br>[CAS Registry Number]<br>InChIKey | $H_s^{cp}$<br>(at $T^{\ominus}$)<br>$\left[\dfrac{\mathrm{mol}}{\mathrm{m^3\,Pa}}\right]$ | $\dfrac{\mathrm{d}\ln H_s^{cp}}{\mathrm{d}(1/T)}$<br><br>[K] | Reference | Type | Note |
|---|---|---|---|---|---|
| $(R)$-1-methyl-4-(1-methylethenyl)-cyclohexene | $3.8\times10^{-4}$ | | Schuhfried et al. (2015) | M | |
| $C_{10}H_{16}$ | $2.6\times10^{-4}$ | | Helburn et al. (2008) | M | |
| $(R$-(+)-limonene; $D$-limonene) | $3.5\times10^{-4}$ | 4500 | Copolovici and Niinemets (2005) | M | |
| [5989-27-5] | $3.8\times10^{-4}$ | | Duchowicz et al. (2020) | V | 186 |
| XMGQYMWWDOXHJM-SNVBAGLBSA-N | $3.9\times10^{-4}$ | | HSDB (2015) | V | |
| | $3.8\times10^{-4}$ | | Mackay et al. (2006a) | V | |
| | $3.8\times10^{-4}$ | | Yaws (2003) | X | 258 |
| | $3.8\times10^{-4}$ | | Yaws (2003) | X | 237 |
| | $7.3\times10^{-4}$ | | Dupeux et al. (2022) | Q | 259 |
| | $6.4\times10^{-4}$ | | Duchowicz et al. (2020) | Q | |
| | $1.2\times10^{-4}$ | | Gharagheizi et al. (2012) | Q | |
| | $3.8\times10^{-4}$ | | Gharagheizi et al. (2010) | Q | 246 |
| | $1.9\times10^{-4}$ | | Modarresi et al. (2007) | Q | 67 |
| | $3.8\times10^{-4}$ | | Yaws (1999) | ? | 21 |
| $(S)$-1-methyl-4-(1-methylethenyl)-cyclohexene | $3.7\times10^{-4}$ | | Schuhfried et al. (2015) | M | |
| $C_{10}H_{16}$ | $3.5\times10^{-4}$ | 4400 | Copolovici and Niinemets (2005) | M | |
| $(S$-(-)-limonene) | | | | | |
| [5989-54-8] | | | | | |
| XMGQYMWWDOXHJM-JTQLQIEISA-N | | | | | |
| 3,7,7-trimethyl-bicyclo[4.1.0]hept-3-ene | $1.6\times10^{-4}$ | | Falk et al. (1990) | M | 14 |
| $C_{10}H_{16}$ | $7.3\times10^{-5}$ | | Copolovici and Niinemets (2005) | V | |
| (3-carene) | $7.3\times10^{-5}$ | | Niinemets and Reichstein (2002) | V | |
| [13466-78-9] | | | | | |
| BQOFWKZOCNGFEC-UHFFFAOYSA-N | | | | | |
| 7-methyl-3-methylene-1,6-octadiene | $4.0\times10^{-4}$ | | Plyasunov and Shock (2000) | L | |
| $C_{10}H_{16}$ | $3.6\times10^{-4}$ | | Schuhfried et al. (2015) | M | |
| (myrcene) | $8.7\times10^{-4}$ | | Fichan et al. (1999) | M | |
| [123-35-3] | $1.5\times10^{-4}$ | | Duchowicz et al. (2020) | V | 186 |
| UAHWPYUMFXYFJY-UHFFFAOYSA-N | $1.1\times10^{-4}$ | | HSDB (2015) | V | |
| | $1.6\times10^{-4}$ | | Copolovici and Niinemets (2005) | V | |
| | $7.2\times10^{-4}$ | 2800 | van Roon et al. (2005) | V | |
| | $1.6\times10^{-4}$ | | Niinemets and Reichstein (2002) | V | |
| | $1.3\times10^{-4}$ | | Dupeux et al. (2022) | Q | 259 |
| | $7.1\times10^{-4}$ | | Duchowicz et al. (2020) | Q | |
| | $1.6\times10^{-4}$ | | Raventos-Duran et al. (2010) | Q | 271, 243 |
| | $6.2\times10^{-5}$ | | Raventos-Duran et al. (2010) | Q | 244 |
| | $2.0\times10^{-5}$ | | Raventos-Duran et al. (2010) | Q | 245 |



Table A2.6: Terpenes and terpenoids (. . . continued)

| Substance<br>Formula<br>(Trivial Name)<br>[CAS Registry Number]<br>InChIKey | $H_s^{cp}$<br>(at $T^\ominus$)<br>$\left[\dfrac{\mathrm{mol}}{\mathrm{m^3\,Pa}}\right]$ | $\dfrac{\mathrm{d\ln} H_s^{cp}}{\mathrm{d}(1/T)}$<br><br>[K] | Reference | Type | Note |
|---|---|---|---|---|---|
| 1-methyl-4-(1-methylethylidene)-<br>cyclohexene | $7.9\times10^{-4}$ | 5500 | Plyasunov and Shock (2000) | L | |
| $C_{10}H_{16}$ | $3.8\times10^{-4}$ | 5300 | Copolovici and Niinemets (2005) | M | |
| ($\alpha$-terpinolene) | $7.0\times10^{-4}$ | | HSDB (2015) | V | |
| [586-62-9] | $3.7\times10^{-4}$ | | Copolovici and Niinemets (2005) | V | |
| MOYAFQVGZZPNRA-UHFFFAOYSA-N | $3.8\times10^{-4}$ | | Niinemets and Reichstein (2002) | V | |
| | $5.7\times10^{-4}$ | 12000 | Li et al. (1998) | V | |
| $\alpha$-phellandrene | $1.8\times10^{-4}$ | | Plyasunov and Shock (2000) | L | |
| $C_{10}H_{16}$ | $1.8\times10^{-4}$ | 4500 | Copolovici and Niinemets (2005) | M | |
| [99-83-2] | $1.8\times10^{-4}$ | | Copolovici and Niinemets (2005) | V | |
| OGLDWXZKYODSOB-UHFFFAOYSA-N | $1.4\times10^{-4}$ | | Niinemets and Reichstein (2002) | V | |
| | $5.3\times10^{-4}$ | | Dupeux et al. (2022) | Q | 259 |
| ($R$)-($-$)-$\alpha$-phellandrene | $3.4\times10^{-4}$ | | Schuhfried et al. (2015) | M | |
| $C_{10}H_{16}$ | | | | | |
| [4221-98-1] | | | | | |
| OGLDWXZKYODSOB-SNVBAGLBSA-N | | | | | |
| $\beta$-phellandrene | $1.8\times10^{-4}$ | 5100 | Copolovici and Niinemets (2005) | M | |
| $C_{10}H_{16}$ | $1.8\times10^{-4}$ | | Copolovici and Niinemets (2005) | V | |
| [555-10-2] | $1.8\times10^{-4}$ | | Niinemets and Reichstein (2002) | V | |
| LFJQCDVYDGGFCH-UHFFFAOYSA-N | $6.1\times10^{-4}$ | | Dupeux et al. (2022) | Q | 259 |
| 3,7-dimethyl-1,3,6-octatriene | $4.0\times10^{-4}$ | | Copolovici and Niinemets (2005) | V | |
| $C_{10}H_{16}$ | | | | | |
| ($\beta$-ocimene) | | | | | |
| [13877-91-3] | | | | | |
| IHPKGUQCSIINRJ-UHFFFAOYSA-N | | | | | |
| ($Z$)-3,7-dimethyl-1,3,6-octatriene | $4.0\times10^{-4}$ | | Niinemets and Reichstein (2002) | V | |
| $C_{10}H_{16}$ | | | | | |
| ($cis$-$\beta$-ocimene) | | | | | |
| [3338-55-4] | | | | | |
| IHPKGUQCSIINRJ-NTMALXAHSA-N | | | | | |
| ($E$)-3,7-dimethyl-1,3,6-octatriene | $3.0\times10^{-4}$ | | Niinemets and Reichstein (2002) | V | |
| $C_{10}H_{16}$ | | | | | |
| ($trans$-$\beta$-ocimene) | | | | | |
| [3779-61-1] | | | | | |
| IHPKGUQCSIINRJ-CSKARUKUSA-N | | | | | |
| 2,2-dimethyl-3-methylene-<br>bicyclo[2.2.1]heptane | $2.3\times10^{-4}$ | | Plyasunov and Shock (2000) | L | |
| $C_{10}H_{16}$ | $1.0\times10^{-4}$ | | HSDB (2015) | V | |
| (camphene) | $3.1\times10^{-4}$ | | Copolovici and Niinemets (2005) | V | |
| [79-92-5] | $6.3\times10^{-4}$ | | Niinemets and Reichstein (2002) | V | |
| CRPUJAZIXJMDBK-UHFFFAOYSA-N | | | | | |





Table A2.6: Terpenes and terpenoids (...continued)

| Substance Formula (Trivial Name) [CAS Registry Number] InChIKey | $H_s^{cp}$ (at $T^\ominus$) $\left[\dfrac{\mathrm{mol}}{\mathrm{m}^3\,\mathrm{Pa}}\right]$ | $\dfrac{\mathrm{d}\ln H_s^{cp}}{\mathrm{d}(1/T)}$ [K] | Reference | Type | Note |
|---|---|---|---|---|---|
| 4-methylene-1-(1-methylethyl)-bicyclo[3.1.0]hexane $C_{10}H_{16}$ (sabinene) [3387-41-5] NDVASEGYNIMXJL-UHFFFAOYSA-N | $1.5\times10^{-4}$ $1.6\times10^{-4}$ $1.6\times10^{-4}$ | | Plyasunov and Shock (2000) Copolovici and Niinemets (2005) Niinemets and Reichstein (2002) | L V V | |
| tricyclo[3.3.1.1(3,7)]decane $C_{10}H_{16}$ (adamantane) [281-23-2] ORILYTVJVMAKLC-UHFFFAOYSA-N | $8.0\times10^{-4}$ $1.1\times10^{-4}$ | 3400 | van Roon et al. (2005) Hilal et al. (2008) | V Q | |
| $\beta$-caryophyllene $C_{15}H_{24}$ [87-44-5] NPNUFJAVOOONJE-IOMPXFEGSA-N | $3.7\times10^{-4}$ $2.6\times10^{-3}$ $6.5\times10^{-4}$ $3.4\times10^{-5}$ $1.0\times10^{-2}$ | 4500 | Copolovici and Niinemets (2015) Dupeux et al. (2022) Wang et al. (2017) Wang et al. (2017) Wang et al. (2017) | M Q Q Q Q | 259 80, 238 80, 239 80, 240 |
| $\alpha$-cedrene $C_{15}H_{24}$ [469-61-4] IRAQOCYXUMOFCW-KYEXWDHISA-N | $2.8\times10^{-4}$ | 4900 | Copolovici and Niinemets (2015) | M | |
| $\alpha$-farnesene $C_{15}H_{24}$ [502-61-4] CXENHBSYCFFKJS-VDQVFBMKSA-N | $3.4\times10^{-4}$ $8.1\times10^{-6}$ | 4300 | Copolovici and Niinemets (2015) Schuhfried et al. (2015) | M M | |
| $\alpha$-humulene $C_{15}H_{24}$ [6753-98-6] FAMPSKZZVDUYOS-HRGUGZIWSA-N | $2.9\times10^{-4}$ | 4700 | Copolovici and Niinemets (2015) | M | |
| $\gamma$-gurjunene $C_{15}H_{24}$ [22567-17-5] DUYRYUZIBGFLDD-UHFFFAOYSA-N | $3.3\times10^{-4}$ | 3700 | Copolovici and Niinemets (2015) | M | |
| isosativene $C_{15}H_{24}$ [24959-83-9] CGZBLYYTRIVVTD-UHFFFAOYSA-N | $3.9\times10^{-4}$ | 3600 | Copolovici and Niinemets (2015) | M | |
| $\alpha$-longipinene $C_{15}H_{24}$ [5989-08-2] HICYDYJTCDBHMZ-UHFFFAOYSA-N | $3.4\times10^{-4}$ | 4100 | Copolovici and Niinemets (2015) | M | |



Table A2.6: Terpenes and terpenoids (... continued)

| Substance Formula (Trivial Name) [CAS Registry Number] InChIKey | $H_s^{cp}$ (at $T^\ominus$) $\left[\dfrac{\text{mol}}{\text{m}^3\,\text{Pa}}\right]$ | $\dfrac{\text{d}\ln H_s^{cp}}{\text{d}(1/T)}$ [K] | Reference | Type | Note |
|---|---|---|---|---|---|
| $\alpha$-neoclovene C$_{15}$H$_{24}$ [4545-68-0] ZCJQJJWNFDNQGZ-UHFFFAOYSA-N | $2.9\times10^{-4}$ | 3100 | Copolovici and Niinemets (2015) | M | |
| $\beta$-neoclovene C$_{15}$H$_{24}$ [56684-96-9] BUDWHMNUSAOQBI-UHFFFAOYSA-N | $2.9\times10^{-4}$ | 4600 | Copolovici and Niinemets (2015) | M | |
| $\gamma$-neoclovene C$_{15}$H$_{24}$ ZYKFQRCKKGFKQR-UHFFFAOYSA-N | $2.9\times10^{-4}$ | 3200 | Copolovici and Niinemets (2015) | M | |
| valencene C$_{15}$H$_{24}$ [4630-07-3] QEBNYNLSCGVZOH-UHFFFAOYSA-N | $3.2\times10^{-4}$ | 4500 | Copolovici and Niinemets (2015) | M | |





### A2.7 Polynuclear aromatics

Table A2.7: Polynuclear aromatics

| Substance / Formula / (Trivial Name) / [CAS Registry Number] / InChIKey | $H_s^{cp}$ (at $T^{\ominus}$) $\left[\dfrac{\mathrm{mol}}{\mathrm{m^3\,Pa}}\right]$ | $\dfrac{\mathrm{d}\ln H_s^{cp}}{\mathrm{d}(1/T)}$ [K] | Reference | Type | Note |
|---|---|---|---|---|---|
| bis(1-methylethyl)-1,1'-biphenyl | $4.5\times10^{-3}$ | | HSDB (2015) | Q | 99 |
| $C_{18}H_{22}$ | $6.4\times10^{-3}$ | | Zhang et al. (2010) | Q | 287, 288 |
| [36876-13-8] | $5.0\times10^{-3}$ | | Zhang et al. (2010) | Q | 287, 289 |
| NUEUMFZLNOCRCQ-UHFFFAOYSA-N | $3.2\times10^{-2}$ | | Zhang et al. (2010) | Q | 287, 290 |
| | $2.0\times10^{-2}$ | | Zhang et al. (2010) | Q | 287, 291 |
| 1,1-bis(3,4-dimethylphenyl)ethane | $1.0\times10^{-2}$ | | Zhang et al. (2010) | Q | 287, 288 |
| $C_{18}H_{22}$ | $1.8\times10^{-2}$ | | Zhang et al. (2010) | Q | 287, 289 |
| [1742-14-9] | $6.5\times10^{-2}$ | | Zhang et al. (2010) | Q | 287, 290 |
| NCSVCMFDHINRJE-UHFFFAOYSA-N | $4.8\times10^{-3}$ | | Zhang et al. (2010) | Q | 287, 291 |
| 1-benzyl-2-(2-methylbenzyl)benzene | $2.1\times10^{-1}$ | | Zhang et al. (2010) | Q | 287, 288 |
| $C_{21}H_{20}$ | $2.5\times10^{-1}$ | | Zhang et al. (2010) | Q | 287, 289 |
| [100404-06-6] | 1.4 | | Zhang et al. (2010) | Q | 287, 290 |
| SMOZJCPUZPHHKQ-UHFFFAOYSA-N | $4.1\times10^{-1}$ | | Zhang et al. (2010) | Q | 287, 291 |
| 2,5-dibenzyltoluene | $2.1\times10^{-1}$ | | Zhang et al. (2010) | Q | 287, 288 |
| $C_{21}H_{20}$ | $2.9\times10^{-1}$ | | Zhang et al. (2010) | Q | 287, 289 |
| [56310-11-3] | 4.5 | | Zhang et al. (2010) | Q | 287, 290 |
| INLIRICAUXXHSB-UHFFFAOYSA-N | $4.1\times10^{-1}$ | | Zhang et al. (2010) | Q | 287, 291 |
| biphenyl | $3.6\times10^{-2}$ | 7000 | Brockbank (2013) | L | 1 |
| $(C_6H_5)_2$ | $3.6\times10^{-2}$ | | Mackay and Shiu (1981) | L | |
| [92-52-4] | $3.4\times10^{-2}$ | | Destaillats and Charles (2002) | M | |
| ZUOUZKKEUPVFJK-UHFFFAOYSA-N | | | Dewulf et al. (1999) | M | 362 |
| | $3.2\times10^{-2}$ | | Shiu and Mackay (1997) | M | |
| | $5.1\times10^{-2}$ | | Fendinger and Glotfelty (1990) | M | |
| | $3.3\times10^{-2}$ | | Mackay and Shiu (1981) | M | |
| | $2.4\times10^{-2}$ | | Mackay et al. (1979) | M | |
| | $3.5\times10^{-2}$ | | Mackay et al. (2006a) | V | |
| | $3.5\times10^{-2}$ | | Mackay et al. (2006b) | V | |
| | $3.6\times10^{-2}$ | | Shiu and Ma (2000) | V | |
| | $3.5\times10^{-2}$ | | Shiu and Mackay (1997) | V | |
| | $3.6\times10^{-2}$ | | Abraham et al. (1994a) | V | |
| | $1.9\times10^{-2}$ | | Mackay et al. (1992a) | V | |
| | $1.2\times10^{-2}$ | | Eastcott et al. (1988) | V | |
| | $1.9\times10^{-2}$ | | Shiu and Mackay (1986) | V | |
| | $7.3\times10^{-2}$ | | Burkhard et al. (1985) | V | |
| | $2.0\times10^{1}$ | 6200 | Bopp (1983) | V | |
| | $3.5\times10^{-2}$ | | Cabani et al. (1981) | V | |
| | $6.4\times10^{-3}$ | | Mackay and Leinonen (1975) | V | |
| | $1.2\times10^{-2}$ | | Bohon and Claussen (1951) | V | |
| | $7.6\times10^{-3}$ | 2900 | Paasivirta et al. (1999) | T | |
| | $1.7\times10^{-2}$ | | Yaws (2003) | X | 258 |
| | $3.9\times10^{-2}$ | | Dupeux et al. (2022) | Q | 259 |



Table A2.7: Polynuclear aromatics (...continued)

| Substance Formula (Trivial Name) [CAS Registry Number] InChIKey | $H_s^{cp}$ (at $T^{\ominus}$) $\left[\dfrac{\text{mol}}{\text{m}^3\,\text{Pa}}\right]$ | $\dfrac{\text{d}\ln H_s^{cp}}{\text{d}(1/T)}$ [K] | Reference | Type | Note |
|---|---|---|---|---|---|
| | $4.5\times10^{-2}$ | | Keshavarz et al. (2022) | Q | |
| | $2.8\times10^{-2}$ | | Duchowicz et al. (2020) | Q | 299 |
| | $4.5\times10^{-2}$ | | Schröder et al. (2010) | Q | 363 |
| | $1.3\times10^{-2}$ | | Hilal et al. (2008) | Q | |
| | $5.0\times10^{-2}$ | | Modarresi et al. (2007) | Q | 67 |
| | | 5100 | Kühne et al. (2005) | Q | |
| | $3.7\times10^{-2}$ | | Yaffe et al. (2003) | Q | 248, 249 |
| | $2.5\times10^{-2}$ | | English and Carroll (2001) | Q | 230, 231 |
| | $4.4\times10^{-4}$ | | Katritzky et al. (1998) | Q | |
| | $8.0\times10^{-3}$ | | Nirmalakhandan and Speece (1988) | Q | |
| | $2.9\times10^{-2}$ | | Arbuckle (1983) | Q | |
| | $3.2\times10^{-2}$ | | Duchowicz et al. (2020) | ? | 185, 21 |
| | | 6000 | Kühne et al. (2005) | ? | |
| | $1.2\times10^{-2}$ | | Yaws and Yang (1992) | ? | 21 |
| 2-methyl-1,1'-biphenyl C$_{13}$H$_{12}$ [643-58-3] ALLIZEAXNXSFGD-UHFFFAOYSA-N | $2.2\times10^{-2}$ $1.0\times10^{-2}$ | | HSDB (2015) Hilal et al. (2008) | Q Q | 99 |
| 3-methyl-1,1'-biphenyl C$_{13}$H$_{12}$ [643-93-6] NPDIDUXTRAITDE-UHFFFAOYSA-N | $1.5\times10^{-2}$ | | Hilal et al. (2008) | Q | |
| 4-methyl-1,1'-biphenyl C$_{13}$H$_{12}$ [644-08-6] ZZLCFHIKESPLTH-UHFFFAOYSA-N | $1.6\times10^{-2}$ | | Hilal et al. (2008) | Q | |
| diphenylmethane C$_{13}$H$_{12}$ (1,1'-methylenebisbenzene) [101-81-5] CZZYITDELCSZES-UHFFFAOYSA-N | $7.7\times10^{-2}$ | | Duchowicz et al. (2020) | V | 186 |
| | $7.6\times10^{-2}$ | | HSDB (2015) | V | |
| | 1.1 | | Mackay et al. (2006a) | V | |
| | 1.1 | | Mackay et al. (1993) | V | |
| | $4.5\times10^{-2}$ | | Meylan and Howard (1991) | V | |
| | $4.7\times10^{-2}$ | | Cabani et al. (1981) | V | |
| | 1.0 | | Mackay et al. (1992b) | X | 364 |
| | $4.3\times10^{-2}$ | | Yaws (2003) | X | 237 |
| | $2.1\times10^{-2}$ | | Duchowicz et al. (2020) | Q | |
| | $5.7\times10^{-3}$ | | Gharagheizi et al. (2010) | Q | 246 |
| | $2.2\times10^{-2}$ | | Hilal et al. (2008) | Q | |
| | $1.6\times10^{-2}$ | | Modarresi et al. (2007) | Q | 67 |
| | $2.1\times10^{-2}$ | | Meylan and Howard (1991) | Q | |
| | $4.3\times10^{-2}$ | | Yaws (1999) | ? | 21 |



Table A2.7: Polynuclear aromatics (...continued)

| Substance Formula (Trivial Name) [CAS Registry Number] InChIKey | $H_s^{cp}$ (at $T^{\ominus}$) $\left[\dfrac{\text{mol}}{\text{m}^3\,\text{Pa}}\right]$ | $\dfrac{\text{d}\ln H_s^{cp}}{\text{d}(1/T)}$ [K] | Reference | Type | Note |
|---|---|---|---|---|---|
| 1,2-diphenylethane $C_{14}H_{14}$ (dibenzyl) [103-29-7] QWUWMCYKGHVNAV-UHFFFAOYSA-N | $5.9\times10^{-2}$ $5.9\times10^{-2}$ $5.9\times10^{-2}$ | | Mackay et al. (2006a) Mackay et al. (1993) Mackay et al. (1992b) | V V X | 364 |
| 3-isopropyl-1,1'-biphenyl $C_{15}H_{16}$ [20282-30-8] LIWRTHVZRZXVFX-UHFFFAOYSA-N | $5.8\times10^{-3}$ | | Ebert et al. (2023) | ? | 365 |
| 4-isopropyl-1,1'-biphenyl $C_{15}H_{16}$ [7116-95-2] KWSHGRJUSUJPQD-UHFFFAOYSA-N | $4.2\times10^{-3}$ | | Ebert et al. (2023) | ? | 365 |
| $o$-terphenyl $C_{18}H_{14}$ [84-15-1] OIAQMFOKAXHPNH-UHFFFAOYSA-N | $1.6\times10^{-1}$ $1.6\times10^{-1}$ $1.3\times10^{-1}$ $3.1\times10^{-1}$ $8.2\times10^{-2}$ $7.3\times10^{-1}$ $4.0$ | | Duchowicz et al. (2020) HSDB (2015) Duchowicz et al. (2020) Zhang et al. (2010) Zhang et al. (2010) Zhang et al. (2010) Zhang et al. (2010) | V V Q Q Q Q Q | 186 287, 288 287, 289 287, 290 287, 291 |
| $m$-terphenyl $C_{18}H_{14}$ [92-06-8] YJTKZCDBKVTVBY-UHFFFAOYSA-N | $2.8$ $2.8$ $1.3\times10^{-1}$ $1.2$ | | Duchowicz et al. (2020) HSDB (2015) Duchowicz et al. (2020) Schröder et al. (2010) | V V Q Q | 186 363 |
| $p$-terphenyl $C_{18}H_{14}$ [92-94-4] XJKSTNDFUHDPQJ-UHFFFAOYSA-N | $6.0$ $2.0\times10^{-2}$ $1.3\times10^{-1}$ $2.9\times10^{-1}$ $3.1\times10^{-1}$ $2.4\times10^{-1}$ $1.1$ $4.0$ $2.5\times10^{-2}$ | | Duchowicz et al. (2020) Mackay et al. (2006a) Yaws (2003) Duchowicz et al. (2020) HSDB (2015) Zhang et al. (2010) Zhang et al. (2010) Zhang et al. (2010) Zhang et al. (2010) Gharagheizi et al. (2010) | V V X Q Q Q Q Q Q | 186 292 237 99 287, 288 287, 289 287, 290 287, 291 246 |
| indene $C_9H_8$ [95-13-6] YBYIRNPNPLQARY-UHFFFAOYSA-N | $6.2\times10^{-3}$ | | HSDB (2015) | Q | 99 |
| 5-ethylidene-2-norbornene $C_9H_{12}$ [16219-75-3] OJOWICOBYCXEKR-KRXBUXKQSA-N | $7.6\times10^{-5}$ | | HSDB (2015) | Q | 99 |





Table A2.7: Polynuclear aromatics (...continued)

| Substance / Formula / (Trivial Name) / [CAS Registry Number] / InChIKey | $H_s^{cp}$ (at $T^\ominus$) $\left[\dfrac{\text{mol}}{\text{m}^3\,\text{Pa}}\right]$ | $\dfrac{\text{d}\ln H_s^{cp}}{\text{d}(1/T)}$ [K] | Reference | Type | Note |
|---|---|---|---|---|---|
| indane | $4.3\times10^{-3}$ | | Plyasunov and Shock (2000) | L | |
| $C_9H_{10}$ | $4.3\times10^{-3}$ | | Mackay et al. (2006a) | V | |
| [496-11-7] | $4.7\times10^{-3}$ | | Abraham et al. (1994a) | V | |
| PQNFLJBBNBOBRQ-UHFFFAOYSA-N | $4.5\times10^{-3}$ | | Yaws (2003) | X | 258 |
| | $4.5\times10^{-3}$ | | Yaws (2003) | X | 237 |
| | $2.4\times10^{-3}$ | | Dupeux et al. (2022) | Q | 259 |
| | $9.6\times10^{-2}$ | | Keshavarz et al. (2022) | Q | |
| | $1.2\times10^{-2}$ | | Duchowicz et al. (2020) | Q | 299 |
| | $4.0\times10^{-3}$ | | Gharagheizi et al. (2012) | Q | |
| | $5.3\times10^{-3}$ | | Gharagheizi et al. (2010) | Q | 246 |
| | $1.2\times10^{-2}$ | | Hilal et al. (2008) | Q | |
| | $2.2\times10^{-3}$ | | Modarresi et al. (2007) | Q | 67 |
| | $4.8\times10^{-3}$ | | Yaffe et al. (2003) | Q | 248, 249 |
| | $7.2\times10^{-3}$ | | English and Carroll (2001) | Q | 230, 231 |
| | $5.8\times10^{-3}$ | | Nirmalakhandan et al. (1997) | Q | |
| | $4.9\times10^{-2}$ | | Duchowicz et al. (2020) | ? | 185, 21 |
| | $4.5\times10^{-3}$ | | Yaws (1999) | ? | 21 |
| azulene | $1.5\times10^{-1}$ | 7800 | Hiatt (2013) | M | |
| $C_{10}H_8$ | | | | | |
| [275-51-4] | | | | | |
| CUFNKYGDVFVPHO-UHFFFAOYSA-N | | | | | |
| naphthalene | $2.1\times10^{-2}$ | 4400 | Schwardt et al. (2021) | L | 1 |
| $C_{10}H_8$ | $2.1\times10^{-2}$ | 5400 | Brockbank (2013) | L | 1 |
| [91-20-3] | $2.1\times10^{-2}$ | | Ma et al. (2010b) | L | 366 |
| UFWIBTONFRDIAS-UHFFFAOYSA-N | $2.2\times10^{-2}$ | | Ma et al. (2010b) | L | 367 |
| | $2.2\times10^{-2}$ | 5300 | Fogg and Sangster (2003) | L | |
| | $2.3\times10^{-2}$ | | Mackay and Shiu (1981) | L | |
| | $3.3\times10^{-2}$ | 6100 | Hiatt (2013) | M | |
| | $6.0\times10^{-2}$ | | Lee et al. (2012) | M | |
| | $4.0\times10^{-2}$ | | Bobadilla et al. (2003) | M | |
| | $2.4\times10^{-2}$ | | Destaillats and Charles (2002) | M | |
| | $1.3\times10^{-2}$ | 3600 | Dewulf et al. (1999) | M | |
| | $1.8\times10^{-2}$ | | Altschuh et al. (1999) | M | |
| | $2.2\times10^{-2}$ | | De Maagd et al. (1998) | M | 12 |
| | $2.2\times10^{-2}$ | | Shiu and Mackay (1997) | M | |
| | $1.7\times10^{-2}$ | 5100 | Kondoh and Nakajima (1997) | M | |
| | $2.3\times10^{-2}$ | 5700 | Alaee et al. (1996) | M | |
| | $2.1\times10^{-2}$ | | Zhang and Pawliszyn (1993) | M | |
| | $1.3\times10^{-2}$ | | Fendinger and Glotfelty (1990) | M | |
| | $2.7\times10^{-2}$ | | Yurteri et al. (1987) | M | 12 |
| | $2.6\times10^{-2}$ | | Webster et al. (1985) | M | |
| | $2.0\times10^{-2}$ | | Mackay et al. (1979) | M | |
| | $1.8\times10^{-2}$ | | Southworth (1979) | M | |
| | $2.2\times10^{-2}$ | 5400 | Schwarz and Wasik (1977) | M | |
| | $2.3\times10^{-2}$ | | Mackay et al. (2006a) | V | |
| | $2.3\times10^{-2}$ | | Shiu and Ma (2000) | V | |





Table A2.7: Polynuclear aromatics (...continued)

| Substance<br>Formula<br>(Trivial Name)<br>[CAS Registry Number]<br>InChIKey | $H_s^{cp}$<br>(at $T^\ominus$)<br>$\left[\dfrac{\mathrm{mol}}{\mathrm{m^3\,Pa}}\right]$ | $\dfrac{\mathrm{d}\ln H_s^{cp}}{\mathrm{d}(1/T)}$<br><br>[K] | Reference | Type | Note |
|---|---|---|---|---|---|
| | $3.2\times10^{-2}$ | | De Maagd et al. (1998) | V | 12 |
| | $2.3\times10^{-2}$ | | Shiu and Mackay (1997) | V | |
| | $2.0\times10^{-2}$ | | Lide and Frederikse (1995) | V | |
| | $2.3\times10^{-2}$ | | Abraham et al. (1994a) | V | |
| | $9.0\times10^{-3}$ | | Hwang et al. (1992) | V | |
| | $7.2\times10^{-3}$ | | Eastcott et al. (1988) | V | |
| | $2.3\times10^{-2}$ | | Cabani et al. (1981) | V | |
| | $2.4\times10^{-2}$ | | Hine and Mookerjee (1975) | V | |
| | $8.4\times10^{-3}$ | | Mackay and Leinonen (1975) | V | |
| | $2.5\times10^{-2}$ | 5100 | Wauchope and Haque (1972) | V | |
| | $2.3\times10^{-2}$ | 5600 | Wauchope and Haque (1972) | V | |
| | $1.9\times10^{-2}$ | | Bohon and Claussen (1951) | V | |
| | $1.1\times10^{-2}$ | 2100 | Paasivirta et al. (1999) | T | |
| | $2.1\times10^{-2}$ | | Mackay et al. (1979) | T | |
| | $7.1\times10^{-3}$ | | Yaws (2003) | X | 258 |
| | $2.1\times10^{-2}$ | 3600 | Goldstein (1982) | X | 298 |
| | $2.7\times10^{-2}$ | | McCarty (1980) | X | 368 |
| | $2.0\times10^{-2}$ | | Smith et al. (1993) | C | |
| | $2.0\times10^{-2}$ | | Ryan et al. (1988) | C | |
| | $1.8\times10^{-2}$ | | Dupeux et al. (2022) | Q | 259 |
| | $1.3\times10^{-1}$ | | Keshavarz et al. (2022) | Q | |
| | $2.4\times10^{-2}$ | | Duchowicz et al. (2020) | Q | 184 |
| | $1.8\times10^{-2}$ | | Parnis et al. (2015) | Q | 369 |
| | $2.7\times10^{-2}$ | | Schröder et al. (2013) | Q | 370 |
| | $1.5\times10^{-2}$ | | Schröder et al. (2010) | Q | 363 |
| | $2.1\times10^{-2}$ | | Hilal et al. (2008) | Q | |
| | $4.0\times10^{-2}$ | | Modarresi et al. (2007) | Q | 67 |
| | | 5200 | Kühne et al. (2005) | Q | |
| | $2.4\times10^{-2}$ | | Yaffe et al. (2003) | Q | 248, 249 |
| | $2.1\times10^{-2}$ | | English and Carroll (2001) | Q | 230, 231 |
| | $3.3\times10^{-4}$ | | Katritzky et al. (1998) | Q | |
| | $5.6\times10^{-2}$ | | Russell et al. (1992) | Q | 279 |
| | $4.3\times10^{-2}$ | | Suzuki et al. (1992) | Q | 232 |
| | $3.2\times10^{-2}$ | | Nirmalakhandan and Speece (1988) | Q | |
| | $3.4\times10^{-2}$ | | Arbuckle (1983) | Q | |
| | $2.2\times10^{-2}$ | | Duchowicz et al. (2020) | ? | 185, 21 |
| | $3.6\times10^{-2}$ | | MacBean (2012a) | ? | |
| | | 5400 | Kühne et al. (2005) | ? | |
| | $8.0\times10^{-3}$ | | Yaws and Yang (1992) | ? | 21 |
| | $2.3\times10^{-2}$ | | Abraham et al. (1990) | ? | |
| naphthalene-d8<br>$C_{10}D_8$<br>[1146-65-2]<br>UFWIBTONFRDIAS-PGRXLJNUSA-N | $3.5\times10^{-2}$ | 5300 | Hiatt (2013) | M | |




Table A2.7: Polynuclear aromatics (...continued)

| Substance Formula (Trivial Name) [CAS Registry Number] InChIKey | $H_s^{cp}$ (at $T^{\ominus}$) $\left[\dfrac{\text{mol}}{\text{m}^3\,\text{Pa}}\right]$ | $\dfrac{\text{d}\ln H_s^{cp}}{\text{d}(1/T)}$ [K] | Reference | Type | Note |
|---|---|---|---|---|---|
| 1-methylnaphthalene | $2.0\times10^{-2}$ | 5800 | Brockbank (2013) | L | 1 |
| $C_{10}H_7CH_3$ | $2.2\times10^{-2}$ | 6100 | Fogg and Sangster (2003) | L | |
| [90-12-0] | $2.2\times10^{-2}$ | | Mackay and Shiu (1981) | L | |
| QPUYECUOLPXSFR-UHFFFAOYSA-N | $4.4\times10^{-2}$ | 5900 | Hiatt (2013) | M | |
| | $1.9\times10^{-2}$ | | Altschuh et al. (1999) | M | |
| | $2.1\times10^{-2}$ | 6100 | Bamford et al. (1999a) | M | |
| | $4.1\times10^{-2}$ | | Shiu and Mackay (1997) | M | |
| | $1.6\times10^{-2}$ | | Fendinger and Glotfelty (1990) | M | |
| | $3.8\times10^{-2}$ | | Mackay and Shiu (1981) | M | |
| | $2.8\times10^{-2}$ | 4900 | Schwarz and Wasik (1977) | M | |
| | $2.2\times10^{-2}$ | | Mackay et al. (2006a) | V | |
| | $2.2\times10^{-2}$ | | Shiu and Ma (2000) | V | |
| | $2.2\times10^{-2}$ | | Shiu and Mackay (1997) | V | |
| | $2.5\times10^{-2}$ | | Abraham et al. (1994a) | V | |
| | $2.5\times10^{-2}$ | | Eastcott et al. (1988) | V | |
| | $2.2\times10^{-2}$ | | Cabani et al. (1981) | V | |
| | $2.2\times10^{-2}$ | | Yaws (2003) | X | 237 |
| | $1.6\times10^{-2}$ | | Keshavarz et al. (2022) | Q | |
| | $1.2\times10^{-2}$ | | Duchowicz et al. (2020) | Q | |
| | $1.6\times10^{-2}$ | | Parnis et al. (2015) | Q | 369 |
| | $2.3\times10^{-2}$ | | Schröder et al. (2013) | Q | 370 |
| | $1.6\times10^{-2}$ | | Gharagheizi et al. (2010) | Q | 246 |
| | $2.8\times10^{-2}$ | | Hilal et al. (2008) | Q | |
| | $3.0\times10^{-2}$ | | Modarresi et al. (2007) | Q | 67 |
| | | 5500 | Kühne et al. (2005) | Q | |
| | $3.8\times10^{-2}$ | | Yaffe et al. (2003) | Q | 248, 249 |
| | $1.5\times10^{-2}$ | | English and Carroll (2001) | Q | 230, 260 |
| | $1.2\times10^{-3}$ | | Katritzky et al. (1998) | Q | |
| | $2.3\times10^{-2}$ | | Nirmalakhandan and Speece (1988) | Q | |
| | $1.9\times10^{-2}$ | | Duchowicz et al. (2020) | ? | 185, 21 |
| | | 5700 | Kühne et al. (2005) | ? | |
| | $2.2\times10^{-2}$ | | Yaws (1999) | ? | 21 |
| | $2.7\times10^{-2}$ | | Yaws and Yang (1992) | ? | 21 |
| 1-methylnaphthalene-d10 | $4.6\times10^{-2}$ | 5400 | Hiatt (2013) | M | |
| $C_{10}D_7CD_3$ | | | | | |
| [38072-94-5] | | | | | |
| QPUYECUOLPXSFR-UZHHFJDZSA-N | | | | | |
| 2-methylnaphthalene | $2.0\times10^{-2}$ | 5100 | Brockbank (2013) | L | 1 |
| $C_{10}H_7CH_3$ | $1.8\times10^{-2}$ | 5600 | Fogg and Sangster (2003) | L | |
| [91-57-6] | $3.5\times10^{-2}$ | 5500 | Hiatt (2013) | M | |
| QIMMUPPBPVKWKM-UHFFFAOYSA-N | $1.6\times10^{-2}$ | | Altschuh et al. (1999) | M | |
| | $1.9\times10^{-2}$ | 5400 | Bamford et al. (1999a) | M | |
| | $2.2\times10^{-2}$ | | De Maagd et al. (1998) | M | 12 |
| | $5.0\times10^{-5}$ | 1200 | Hansen et al. (1993) | M | 281 |
| | $3.1\times10^{-2}$ | | Fendinger and Glotfelty (1990) | M | |
| | $2.0\times10^{-2}$ | | Mackay et al. (2006a) | V | |



Table A2.7: Polynuclear aromatics (...continued)

| Substance Formula (Trivial Name) [CAS Registry Number] InChIKey | $H_s^{cp}$ (at $T^\ominus$) $\left[\dfrac{\mathrm{mol}}{\mathrm{m}^3\,\mathrm{Pa}}\right]$ | $\dfrac{\mathrm{d}\ln H_s^{cp}}{\mathrm{d}(1/T)}$ [K] | Reference | Type | Note |
|---|---|---|---|---|---|
| | $2.6\times10^{-2}$ | | De Maagd et al. (1998) | V | 12 |
| | $2.0\times10^{-2}$ | | Shiu and Mackay (1997) | V | |
| | $2.4\times10^{-2}$ | | Meylan and Howard (1991) | V | |
| | $2.0\times10^{-2}$ | | Eastcott et al. (1988) | V | |
| | $2.4\times10^{-2}$ | | Mackay and Shiu (1981) | V | |
| | $2.0\times10^{-2}$ | | Mackay et al. (1992b) | X | 364 |
| | $2.3\times10^{-2}$ | | Yaws (2003) | X | 237 |
| | $1.6\times10^{-2}$ | | Keshavarz et al. (2022) | Q | |
| | $1.2\times10^{-2}$ | | Duchowicz et al. (2020) | Q | |
| | $1.6\times10^{-2}$ | | Parnis et al. (2015) | Q | 369 |
| | $2.2\times10^{-2}$ | | Schröder et al. (2013) | Q | 370 |
| | $1.6\times10^{-2}$ | | Gharagheizi et al. (2010) | Q | 246 |
| | $2.6\times10^{-2}$ | | Hilal et al. (2008) | Q | |
| | $2.4\times10^{-2}$ | | Modarresi et al. (2007) | Q | 67 |
| | | 5500 | Kühne et al. (2005) | Q | |
| | $2.4\times10^{-2}$ | | Yaffe et al. (2003) | Q | 248, 249 |
| | $1.2\times10^{-3}$ | | Katritzky et al. (1998) | Q | |
| | $1.7\times10^{-2}$ | | Meylan and Howard (1991) | Q | |
| | $1.9\times10^{-2}$ | | Duchowicz et al. (2020) | ? | 185, 21 |
| | | 5700 | Kühne et al. (2005) | ? | |
| | $2.3\times10^{-2}$ | | Yaws (1999) | ? | 21 |
| | $2.0\times10^{-2}$ | | Yaws and Yang (1992) | ? | 21 |
| | | | Shiu and Ma (2000) | W | 360 |
| 1-ethylnaphthalene $C_{10}H_7C_2H_5$ [1127-76-0] ZMXIYERNXPIYFR-UHFFFAOYSA-N | $2.0\times10^{-2}$ | 5900 | Brockbank (2013) | L | 1 |
| | $2.6\times10^{-2}$ | | Mackay and Shiu (1981) | L | |
| | $1.4\times10^{-2}$ | | Altschuh et al. (1999) | M | |
| | $2.2\times10^{-2}$ | 4800 | Schwarz and Wasik (1977) | M | |
| | $2.6\times10^{-2}$ | | Mackay et al. (2006a) | V | |
| | $2.7\times10^{-2}$ | | Eastcott et al. (1988) | V | |
| | $2.3\times10^{-2}$ | | Cabani et al. (1981) | V | |
| | $2.6\times10^{-2}$ | | Mackay et al. (1992b) | X | 364 |
| | $2.0\times10^{-2}$ | | Yaws (2003) | X | 237 |
| | $2.2\times10^{-2}$ | | Keshavarz et al. (2022) | Q | |
| | $1.2\times10^{-2}$ | | Duchowicz et al. (2020) | Q | 184 |
| | $1.9\times10^{-2}$ | | Schröder et al. (2013) | Q | 370 |
| | $1.6\times10^{-2}$ | | Gharagheizi et al. (2010) | Q | 246 |
| | $2.8\times10^{-2}$ | | Hilal et al. (2008) | Q | |
| | $2.2\times10^{-2}$ | | Modarresi et al. (2007) | Q | 67 |
| | $4.8\times10^{-3}$ | | Yaffe et al. (2003) | Q | 248, 249 |
| | $9.5\times10^{-3}$ | | English and Carroll (2001) | Q | 230, 231 |
| | $2.2\times10^{-1}$ | | Nirmalakhandan et al. (1997) | Q | |
| | $1.4\times10^{-2}$ | | Duchowicz et al. (2020) | ? | 185, 21 |
| | $2.0\times10^{-2}$ | | Yaws (1999) | ? | 21 |
| | $2.7\times10^{-2}$ | | Yaws and Yang (1992) | ? | 21 |



Table A2.7: Polynuclear aromatics (. . . continued)

| Substance Formula (Trivial Name) [CAS Registry Number] InChIKey | $H_s^{cp}$ (at $T^{\ominus}$) $\left[\dfrac{\mathrm{mol}}{\mathrm{m^3\,Pa}}\right]$ | $\dfrac{\mathrm{d}\ln H_s^{cp}}{\mathrm{d}(1/T)}$ [K] | Reference | Type | Note |
|---|---|---|---|---|---|
| 2-ethylnaphthalene | $1.2\times10^{-2}$ | | Mackay and Shiu (1981) | L | |
| $C_{10}H_7C_2H_5$ | $1.8\times10^{-2}$ | | Altschuh et al. (1999) | M | |
| [939-27-5] | $1.3\times10^{-2}$ | | Mackay et al. (2006a) | V | |
| RJTJVVYSTUQWNI-UHFFFAOYSA-N | $1.6\times10^{-2}$ | | Eastcott et al. (1988) | V | |
| | $1.3\times10^{-2}$ | | Mackay et al. (1992b) | X | 364 |
| | $8.8\times10^{-3}$ | | Yaws (2003) | X | 237 |
| | $2.2\times10^{-2}$ | | Keshavarz et al. (2022) | Q | |
| | $1.2\times10^{-2}$ | | Duchowicz et al. (2020) | Q | 299 |
| | $1.9\times10^{-2}$ | | Schröder et al. (2013) | Q | 370 |
| | $3.6\times10^{-2}$ | | Gharagheizi et al. (2012) | Q | |
| | $1.6\times10^{-2}$ | | Gharagheizi et al. (2010) | Q | 246 |
| | $1.9\times10^{-2}$ | | Hilal et al. (2008) | Q | |
| | $2.2\times10^{-2}$ | | Modarresi et al. (2007) | Q | 67 |
| | $1.2\times10^{-2}$ | | Yaffe et al. (2003) | Q | 248, 249 |
| | $1.8\times10^{-2}$ | | Duchowicz et al. (2020) | ? | 185, 21 |
| | $1.6\times10^{-2}$ | | Yaws and Yang (1992) | ? | 21 |
| 1,2,3,4-tetrahydronaphthalene | $6.4\times10^{-3}$ | 5300 | Plyasunov and Shock (2000) | L | |
| $C_{10}H_{12}$ | $5.1\times10^{-3}$ | 5400 | Ashworth et al. (1988) | M | 278 |
| (tetralin) | $7.3\times10^{-3}$ | | Duchowicz et al. (2020) | V | 186 |
| [119-64-2] | $2.1\times10^{-3}$ | | Mackay et al. (1993) | V | |
| CXWXQJXEFPUFDZ-UHFFFAOYSA-N | $5.8\times10^{-3}$ | | Yaws (2003) | X | 237, 297 |
| | $1.2\times10^{-2}$ | | Duchowicz et al. (2020) | Q | |
| | $5.8\times10^{-3}$ | | HSDB (2015) | Q | 99 |
| | $3.6\times10^{-3}$ | | Gharagheizi et al. (2012) | Q | |
| | $4.3\times10^{-3}$ | | Gharagheizi et al. (2010) | Q | 246 |
| | $1.2\times10^{-2}$ | | Hilal et al. (2008) | Q | |
| | $4.3\times10^{-3}$ | | Modarresi et al. (2007) | Q | 67 |
| | | 4900 | Kühne et al. (2005) | Q | |
| | | 5300 | Kühne et al. (2005) | ? | |
| | $7.2\times10^{-3}$ | | Yaws (1999) | ? | 21, 297 |
| 1,2-dimethylnaphthalene | $4.8\times10^{-2}$ | | Yaws (2003) | X | 258 |
| $C_{12}H_{12}$ | $4.8\times10^{-2}$ | | Yaws (2003) | X | 237 |
| [573-98-8] | $2.6\times10^{-2}$ | | Dupeux et al. (2022) | Q | 259 |
| QNLZIZAQLLYXTC-UHFFFAOYSA-N | $4.2\times10^{-2}$ | | Schröder et al. (2013) | Q | 370 |
| | $1.3\times10^{-2}$ | | Gharagheizi et al. (2010) | Q | 246 |
| 1,3-dimethylnaphthalene | $1.7\times10^{-2}$ | | Duchowicz et al. (2020) | V | 186 |
| $C_{12}H_{12}$ | $2.6\times10^{-2}$ | | Cabani et al. (1981) | V | |
| [575-41-7] | $6.1\times10^{-3}$ | | Duchowicz et al. (2020) | Q | |
| QHJMFSMPSZREIF-UHFFFAOYSA-N | $2.0\times10^{-2}$ | | Schröder et al. (2013) | Q | 370 |
| | $2.8\times10^{-2}$ | | Gharagheizi et al. (2012) | Q | |
| | $2.9\times10^{-2}$ | | Hilal et al. (2008) | Q | |
| | $4.6\times10^{-2}$ | | Yaffe et al. (2003) | Q | 248, 272 |
| | $1.1\times10^{-2}$ | | English and Carroll (2001) | Q | 230, 231 |
| | $1.9\times10^{-1}$ | | Nirmalakhandan et al. (1997) | Q | |
| | $1.4\times10^{-2}$ | | Yaws and Yang (1992) | ? | 21 |



Table A2.7: Polynuclear aromatics (...continued)

| Substance Formula (Trivial Name) [CAS Registry Number] InChIKey | $H_s^{cp}$ (at $T^\ominus$) $\left[\dfrac{\mathrm{mol}}{\mathrm{m^3\,Pa}}\right]$ | $\dfrac{\mathrm{d}\ln H_s^{cp}}{\mathrm{d}(1/T)}$ [K] | Reference | Type | Note |
|---|---|---|---|---|---|
| 1,4-dimethylnaphthalene $C_{12}H_{12}$ [571-58-4] APQSQLNWAIULLK-UHFFFAOYSA-N | $2.6\times10^{-2}$ | | Duchowicz et al. (2020) | V | 186 |
| | $3.2\times10^{-2}$ | | Mackay et al. (2006a) | V | |
| | $4.7\times10^{-2}$ | | Cabani et al. (1981) | V | |
| | $6.1\times10^{-3}$ | | Duchowicz et al. (2020) | Q | |
| | $2.0\times10^{-2}$ | | Schröder et al. (2013) | Q | 370 |
| | $2.9\times10^{-2}$ | | Gharagheizi et al. (2012) | Q | |
| | $4.4\times10^{-2}$ | | Hilal et al. (2008) | Q | |
| | $3.7\times10^{-2}$ | | Yaffe et al. (2003) | Q | 248, 272 |
| | $1.9\times10^{-1}$ | | Nirmalakhandan et al. (1997) | Q | |
| | $1.3\times10^{-2}$ | | Maniere et al. (2011) | ? | 165 |
| | $2.0\times10^{-2}$ | | Yaws and Yang (1992) | ? | 21 |
| 1,5-dimethylnaphthalene $C_{12}H_{12}$ [571-61-9] SDDBCEWUYXVGCQ-UHFFFAOYSA-N | $2.8\times10^{-2}$ | | Shiu and Mackay (1997) | M | |
| | $1.3\times10^{-2}$ | | Yaws (2003) | X | 237 |
| | $2.2\times10^{-2}$ | | Keshavarz et al. (2022) | Q | |
| | $6.1\times10^{-3}$ | | Duchowicz et al. (2020) | Q | 184 |
| | $1.4\times10^{-2}$ | | Parnis et al. (2015) | Q | 369 |
| | $2.1\times10^{-2}$ | | Schröder et al. (2013) | Q | 370 |
| | $2.7\times10^{-2}$ | | Gharagheizi et al. (2012) | Q | |
| | $1.3\times10^{-2}$ | | Gharagheizi et al. (2010) | Q | 246 |
| | $3.3\times10^{-2}$ | | Hilal et al. (2008) | Q | |
| | $2.4\times10^{-2}$ | | Modarresi et al. (2007) | Q | 67 |
| | $2.9\times10^{-2}$ | | Yaffe et al. (2003) | Q | 248, 249 |
| | $1.1\times10^{-1}$ | | Nirmalakhandan and Speece (1988) | Q | |
| | $2.8\times10^{-2}$ | | Duchowicz et al. (2020) | ? | 185, 21 |
| | $1.6\times10^{-2}$ | | Yaws and Yang (1992) | ? | 21 |
| 1,6-dimethylnaphthalene $C_{12}H_{12}$ [575-43-9] CBMXCNPQDUJNHT-UHFFFAOYSA-N | $2.3\times10^{-2}$ | | HSDB (2015) | Q | 99 |
| | $3.8\times10^{-2}$ | | Schröder et al. (2013) | Q | 370 |
| 1,7-dimethylnaphthalene $C_{12}H_{12}$ [575-37-1] SPUWFVKLHHEKGV-UHFFFAOYSA-N | $3.9\times10^{-2}$ | | Ebert et al. (2023) | ? | 371 |
| 1,8-dimethylnaphthalene $C_{12}H_{12}$ [569-41-5] XAABPYINPXYOLM-UHFFFAOYSA-N | $2.6\times10^{-2}$ | | Schröder et al. (2013) | Q | 370 |
| 2,3-dimethylnaphthalene $C_{12}H_{12}$ [581-40-8] WWGUMAYGTYQSGA-UHFFFAOYSA-N | $1.6\times10^{-2}$ | | Mackay et al. (2006a) | V | |
| | $6.4\times10^{-2}$ | | Eastcott et al. (1988) | V | |
| | $4.4\times10^{-2}$ | | Cabani et al. (1981) | V | |
| | $1.3\times10^{-2}$ | | Yaws (2003) | X | 237 |
| | $1.1\times10^{-2}$ | | Meylan and Howard (1991) | C | |
| | $2.2\times10^{-2}$ | | Keshavarz et al. (2022) | Q | |
| | $6.1\times10^{-3}$ | | Duchowicz et al. (2020) | Q | 299 |
| | $2.5\times10^{-2}$ | | Schröder et al. (2013) | Q | 370 |





Table A2.7: Polynuclear aromatics (...continued)

| Substance Formula (Trivial Name) [CAS Registry Number] InChIKey | $H_s^{cp}$ (at $T^\ominus$) $\left[\dfrac{\mathrm{mol}}{\mathrm{m}^3\,\mathrm{Pa}}\right]$ | $\dfrac{\mathrm{d}\ln H_s^{cp}}{\mathrm{d}(1/T)}$ [K] | Reference | Type | Note |
|---|---|---|---|---|---|
| | $3.0\times10^{-2}$ | | Gharagheizi et al. (2012) | Q | |
| | $1.3\times10^{-2}$ | | Gharagheizi et al. (2010) | Q | 246 |
| | $3.6\times10^{-2}$ | | Hilal et al. (2008) | Q | |
| | $2.0\times10^{-2}$ | | Modarresi et al. (2007) | Q | 67 |
| | $4.6\times10^{-2}$ | | Yaffe et al. (2003) | Q | 248, 249 |
| | $1.3\times10^{-1}$ | | Nirmalakhandan et al. (1997) | Q | |
| | $1.5\times10^{-2}$ | | Meylan and Howard (1991) | Q | |
| | $1.1\times10^{-2}$ | | Duchowicz et al. (2020) | ? | 185, 21 |
| | $1.7\times10^{-2}$ | | Yaws and Yang (1992) | ? | 21 |
| 2,6-dimethylnaphthalene C$_{12}$H$_{12}$ [581-42-0] YGYNBBAUIYTWBF-UHFFFAOYSA-N | $7.8\times10^{-3}$ | | Mackay et al. (2006a) | V | |
| | $6.2\times10^{-2}$ | | Eastcott et al. (1988) | V | |
| | $3.4\times10^{-2}$ | | Cabani et al. (1981) | V | |
| | $2.0\times10^{-2}$ | | Schröder et al. (2013) | Q | 370 |
| | $2.8\times10^{-2}$ | | Gharagheizi et al. (2012) | Q | |
| | $3.2\times10^{-2}$ | | Hilal et al. (2008) | Q | |
| | $2.2\times10^{-2}$ | | Modarresi et al. (2007) | Q | 67 |
| | $3.7\times10^{-2}$ | | Yaffe et al. (2003) | Q | 248, 249 |
| | $1.9\times10^{-1}$ | | Nirmalakhandan et al. (1997) | Q | |
| | $8.2\times10^{-3}$ | | Yaws and Yang (1992) | ? | 21 |
| 2,7-dimethylnaphthalene C$_{12}$H$_{12}$ [582-16-1] LRQYSMQNJLZKPS-UHFFFAOYSA-N | $2.1\times10^{-2}$ | | Schröder et al. (2013) | Q | 370 |
| acenaphthene C$_{12}$H$_{10}$ [83-32-9] CWRYPZZKDGJXCA-UHFFFAOYSA-N | $5.4\times10^{-2}$ | 6500 | Schwardt et al. (2021) | L | 1 |
| | $7.2\times10^{-2}$ | 5400 | Brockbank (2013) | L | 1 |
| | $7.2\times10^{-2}$ | | Ma et al. (2010b) | L | 366 |
| | $7.0\times10^{-2}$ | | Ma et al. (2010b) | L | 367 |
| | $5.5\times10^{-2}$ | 6500 | Fogg and Sangster (2003) | L | |
| | $4.2\times10^{-2}$ | | Mackay and Shiu (1981) | L | |
| | $2.6\times10^{-1}$ | | Lee et al. (2012) | M | |
| | $5.4\times10^{-2}$ | 6600 | Bamford et al. (1999a) | M | |
| | $6.2\times10^{-2}$ | | Shiu and Mackay (1997) | M | |
| | $1.1\times10^{-1}$ | | Zhang and Pawliszyn (1993) | M | |
| | $1.6\times10^{-1}$ | | Fendinger and Glotfelty (1990) | M | |
| | $6.4\times10^{-3}$ | | Mackay and Shiu (1981) | M | |
| | $4.1\times10^{-2}$ | | Warner et al. (1980) | M | |
| | $6.8\times10^{-2}$ | | Mackay et al. (1979) | M | |
| | $8.2\times10^{-2}$ | | Mackay et al. (2006a) | V | |
| | $8.2\times10^{-2}$ | | Shiu and Ma (2000) | V | |
| | $8.2\times10^{-2}$ | | Shiu and Mackay (1997) | V | |
| | $1.2\times10^{-2}$ | | Hwang et al. (1992) | V | |
| | $9.5\times10^{-2}$ | | Eastcott et al. (1988) | V | |
| | $8.2\times10^{-2}$ | | Cabani et al. (1981) | V | |
| | $1.2\times10^{-1}$ | | Hine and Mookerjee (1975) | V | |
| | $1.2\times10^{-1}$ | 6000 | Wauchope and Haque (1972) | V | |
| | $3.4\times10^{-2}$ | 2900 | Paasivirta et al. (1999) | T | |



Table A2.7: Polynuclear aromatics (...continued)

| Substance<br>Formula<br>(Trivial Name)<br>[CAS Registry Number]<br>InChIKey | $H_s^{cp}$<br>(at $T^\ominus$)<br>$\left[\dfrac{\mathrm{mol}}{\mathrm{m^3\,Pa}}\right]$ | $\dfrac{\mathrm{d}\ln H_s^{cp}}{\mathrm{d}(1/T)}$<br><br>[K] | Reference | Type | Note |
|---|---|---|---|---|---|
| | $4.1\times10^{-2}$ | 2800 | Goldstein (1982) | X | 298 |
| | $5.2\times10^{-2}$ | | McCarty (1980) | X | 368 |
| | $6.4\times10^{-2}$ | | HSDB (2015) | C | |
| | $4.1\times10^{-2}$ | | Smith et al. (1993) | C | |
| | $4.0\times10^{-2}$ | | Ryan et al. (1988) | C | |
| | $4.1\times10^{-2}$ | | Shen (1982) | C | |
| | $2.4\times10^{-1}$ | | Keshavarz et al. (2022) | Q | |
| | $4.8\times10^{-2}$ | | Duchowicz et al. (2020) | Q | |
| | $9.2\times10^{-2}$ | | Abraham et al. (2019) | Q | |
| | $3.4\times10^{-2}$ | | Parnis et al. (2015) | Q | 369 |
| | $9.2\times10^{-2}$ | | Gharagheizi et al. (2012) | Q | |
| | $2.9\times10^{-2}$ | | Schröder et al. (2010) | Q | 363 |
| | $2.2\times10^{-1}$ | | Hilal et al. (2008) | Q | |
| | $4.4\times10^{-2}$ | | Modarresi et al. (2007) | Q | 67 |
| | | 5500 | Kühne et al. (2005) | Q | |
| | $6.7\times10^{-2}$ | | Yaffe et al. (2003) | Q | 248, 249 |
| | $9.9\times10^{-2}$ | | Suzuki et al. (1992) | Q | 232 |
| | $1.1\times10^{-1}$ | | Nirmalakhandan and Speece (1988) | Q | |
| | $7.9\times10^{-2}$ | | Arbuckle (1983) | Q | |
| | $5.4\times10^{-2}$ | | Duchowicz et al. (2020) | ? | 185, 21 |
| | | 6600 | Kühne et al. (2005) | ? | |
| acenaphthylene<br>$C_{12}H_8$<br>[208-96-8]<br>HXGDTGSAIMULJN-UHFFFAOYSA-N | $8.5\times10^{-2}$ | 6300 | Brockbank (2013) | L | 1 |
| | $8.2\times10^{-2}$ | | Ma et al. (2010b) | L | 366 |
| | $1.0\times10^{-1}$ | | Ma et al. (2010b) | L | 367 |
| | $9.1\times10^{-2}$ | 6700 | Fogg and Sangster (2003) | L | |
| | $7.9\times10^{-2}$ | 6600 | Bamford et al. (1999a) | M | |
| | $8.8\times10^{-2}$ | | Fendinger and Glotfelty (1990) | M | |
| | $8.7\times10^{-2}$ | | Warner et al. (1980) | M | |
| | $8.7\times10^{-1}$ | | HSDB (2015) | V | |
| | $1.2\times10^{-1}$ | | Mackay et al. (2006a) | V | |
| | $1.2\times10^{-1}$ | | Shiu and Mackay (1997) | V | |
| | $1.2\times10^{-1}$ | 5000 | Paasivirta et al. (1999) | T | |
| | $8.7\times10^{-2}$ | | Smith et al. (1993) | C | |
| | $8.4\times10^{-2}$ | | Ryan et al. (1988) | C | |
| | $8.7\times10^{-2}$ | | Shen (1982) | C | |
| | $2.4\times10^{-1}$ | | Keshavarz et al. (2022) | Q | |
| | $5.7\times10^{-2}$ | | Duchowicz et al. (2020) | Q | 184 |
| | $6.2\times10^{-2}$ | | Parnis et al. (2015) | Q | 369 |
| | $1.1\times10^{-1}$ | | Hilal et al. (2008) | Q | |
| | $2.2\times10^{-2}$ | | Modarresi et al. (2007) | Q | 67 |
| | | 5600 | Kühne et al. (2005) | Q | |
| | $8.6\times10^{-2}$ | | Yaffe et al. (2003) | Q | 248, 249 |
| | $8.7\times10^{-2}$ | | Duchowicz et al. (2020) | ? | 185, 21 |
| | | 6600 | Kühne et al. (2005) | ? | |
| | | | Shiu and Ma (2000) | W | 360 |





Table A2.7: Polynuclear aromatics (...continued)

| Substance Formula (Trivial Name) [CAS Registry Number] InChIKey | $H_s^{cp}$ (at $T^\ominus$) $\left[\dfrac{\mathrm{mol}}{\mathrm{m^3\,Pa}}\right]$ | $\dfrac{\mathrm{d}\ln H_s^{cp}}{\mathrm{d}(1/T)}$ [K] | Reference | Type | Note |
|---|---|---|---|---|---|
| 1,2,4-trimethylnaphthalene C$_{13}$H$_{14}$ [2717-42-2] JCNGSJUYPCVGAM-UHFFFAOYSA-N | $1.3\times10^{-2}$ | | Parnis et al. (2015) | Q | 369 |
| 1,3,7-trimethylnaphthalene C$_{13}$H$_{14}$ [2131-38-6] HXDVFWJRDVUZFT-UHFFFAOYSA-N | $6.0\times10^{-2}$ | | Ebert et al. (2023) | ? | 371 |
| 1,3,8-trimethylnaphthalene C$_{13}$H$_{14}$ [17057-91-9] XYTKCJHHXQVFCK-UHFFFAOYSA-N | $6.7\times10^{-2}$ | | Ebert et al. (2023) | ? | 371 |
| 1,4,5-trimethylnaphthalene C$_{13}$H$_{14}$ [2131-41-1] FSAWRQYDMHSDRN-UHFFFAOYSA-N | $1.8\times10^{-2}$ $4.3\times10^{-2}$ $2.3\times10^{-2}$ | | Mackay et al. (2006a) Eastcott et al. (1988) Schröder et al. (2013) | V V Q | 370 |
| 1,4,6-trimethylnaphthalene C$_{13}$H$_{14}$ [2131-42-2] VGKRZAKNKJAKDN-UHFFFAOYSA-N | $4.3\times10^{-2}$ | | Ebert et al. (2023) | ? | 371 |
| 2,3,6-trimethylnaphthalene C$_{13}$H$_{14}$ [829-26-5] UNBZRJCHIWTUHB-UHFFFAOYSA-N | $2.8\times10^{-2}$ | | Ebert et al. (2023) | ? | 371 |
| 1-(1-methylethyl)naphthalene C$_{13}$H$_{14}$ (1-isopropylnaphthalene) [6158-45-8] PMPBFICDXLLSRM-UHFFFAOYSA-N | $3.3\times10^{-2}$ | | Schröder et al. (2013) | Q | 370 |
| 2-(1-methylethyl)naphthalene C$_{13}$H$_{14}$ (2-isopropylnaphthalene) [2027-17-0] TVYVQNHYIHAJTD-UHFFFAOYSA-N | $1.2\times10^{-2}$ $3.1\times10^{-2}$ | | HSDB (2015) Schröder et al. (2013) | Q Q | 99 370 |
| 2-ethyl-6-methylnaphthalene C$_{13}$H$_{14}$ [7372-86-3] ZOYUJOHRFWIQTH-UHFFFAOYSA-N | $1.2\times10^{-2}$ | | Parnis et al. (2015) | Q | 369 |



Table A2.7: Polynuclear aromatics (...continued)

| Substance Formula (Trivial Name) [CAS Registry Number] InChIKey | $H_s^{cp}$ (at $T^{\ominus}$) $\left[\dfrac{\mathrm{mol}}{\mathrm{m^3\,Pa}}\right]$ | $\dfrac{\mathrm{d\ln}H_s^{cp}}{\mathrm{d}(1/T)}$ [K] | Reference | Type | Note |
|---|---|---|---|---|---|
| 2,3-benzindene | $1.3\times10^{-1}$ | 6100 | Brockbank (2013) | L | 1 |
| $C_{13}H_{10}$ | $1.1\times10^{-1}$ | | Ma et al. (2010b) | L | 366 |
| (fluorene) | $1.1\times10^{-1}$ | | Ma et al. (2010b) | L | 367 |
| [86-73-7] | $1.1\times10^{-1}$ | 6000 | Fogg and Sangster (2003) | L | |
| NIHNNTQXNPWCJQ-UHFFFAOYSA-N | $1.2\times10^{-1}$ | | Mackay and Shiu (1981) | L | |
| | $3.2\times10^{-1}$ | | Lee et al. (2012) | M | |
| | $1.0\times10^{-1}$ | 6200 | Bamford et al. (1999a) | M | |
| | $7.9\times10^{-2}$ | 7400 | Bamford et al. (1999b) | M | |
| | $1.5\times10^{-1}$ | | De Maagd et al. (1998) | M | 12 |
| | $1.0\times10^{-1}$ | | Shiu and Mackay (1997) | M | |
| | $1.6\times10^{-1}$ | | Fendinger and Glotfelty (1990) | M | |
| | $9.9\times10^{-2}$ | | Mackay and Shiu (1981) | M | |
| | $8.4\times10^{-2}$ | | Warner et al. (1980) | M | |
| | $1.3\times10^{-1}$ | | Mackay et al. (2006a) | V | |
| | $1.3\times10^{-1}$ | | Shiu and Ma (2000) | V | |
| | $1.7\times10^{-1}$ | | De Maagd et al. (1998) | V | 12 |
| | $1.3\times10^{-1}$ | | Shiu and Mackay (1997) | V | |
| | $1.5\times10^{-2}$ | | Hwang et al. (1992) | V | |
| | $1.1\times10^{-1}$ | | Eastcott et al. (1988) | V | |
| | $1.3\times10^{-1}$ | | Cabani et al. (1981) | V | |
| | $1.3\times10^{-1}$ | 6400 | Wauchope and Haque (1972) | V | |
| | $2.3\times10^{-2}$ | 3700 | Paasivirta et al. (1999) | T | |
| | $8.4\times10^{-2}$ | 3000 | Goldstein (1982) | X | 298 |
| | $4.7\times10^{-2}$ | | McCarty (1980) | X | 368 |
| | $9.9\times10^{-2}$ | | HSDB (2015) | C | |
| | $8.4\times10^{-2}$ | | Smith et al. (1993) | C | |
| | $8.4\times10^{-2}$ | | Ryan et al. (1988) | C | |
| | $8.4\times10^{-2}$ | | Shen (1982) | C | |
| | $3.2\times10^{-1}$ | | Keshavarz et al. (2022) | Q | |
| | $5.7\times10^{-2}$ | | Duchowicz et al. (2020) | Q | 299 |
| | $1.1\times10^{-1}$ | | Abraham et al. (2019) | Q | |
| | $1.0\times10^{-1}$ | | Parnis et al. (2015) | Q | 369 |
| | $9.0\times10^{-2}$ | | Schröder et al. (2010) | Q | 363 |
| | $9.2\times10^{-2}$ | | Hilal et al. (2008) | Q | |
| | $4.1\times10^{-2}$ | | Modarresi et al. (2007) | Q | 67 |
| | | 5100 | Kühne et al. (2005) | Q | |
| | $1.2\times10^{-1}$ | | Yaffe et al. (2003) | Q | 248, 249 |
| | $5.4\times10^{-2}$ | | English and Carroll (2001) | Q | 230, 274 |
| | $2.0\times10^{-1}$ | | Nirmalakhandan and Speece (1988) | Q | |
| | $1.0\times10^{-1}$ | | Duchowicz et al. (2020) | ? | 185, 21 |
| | | 5400 | Kühne et al. (2005) | ? | |
| | $1.2\times10^{-1}$ | | Abraham et al. (1990) | ? | |
| 1-methyl-9H-fluorene | $1.5\times10^{-1}$ | | Duchowicz et al. (2020) | V | 186 |
| $C_{14}H_{12}$ | $2.9\times10^{-2}$ | | Duchowicz et al. (2020) | Q | |
| [1730-37-6] | | | | | |
| GKEUODMJRFDLJY-UHFFFAOYSA-N | | | | | |



Table A2.7: Polynuclear aromatics (. . . continued)

| Substance Formula (Trivial Name) [CAS Registry Number] InChIKey | $H_s^{cp}$ (at $T^{\ominus}$) $\left[\dfrac{\text{mol}}{\text{m}^3\,\text{Pa}}\right]$ | $\dfrac{\text{d}\ln H_s^{cp}}{\text{d}(1/T)}$ [K] | Reference | Type | Note |
|---|---|---|---|---|---|
| 2-methyl-9H-fluorene $C_{14}H_{12}$ [1430-97-3] RKJHJMAZNPASHY-UHFFFAOYSA-N | $8.6\times10^{-2}$ | | Parnis et al. (2015) | Q | 369 |
| 9-methyl-9H-fluorene $C_{14}H_{12}$ [2523-37-7] ZVEJRZRAUYJYCO-UHFFFAOYSA-N | $9.1\times10^{-2}$ | | Parnis et al. (2015) | Q | 369 |
| 1,2,5,6-tetramethylnaphthalene $C_{14}H_{16}$ [2131-43-3] ONIJFQFZCKJNDH-UHFFFAOYSA-N | $1.3\times10^{-2}$ | | Parnis et al. (2015) | Q | 369 |
| 1,4,6,7-tetramethylnaphthalene $C_{14}H_{16}$ [13764-18-6] VPSPONOBLZCLIU-UHFFFAOYSA-N | $8.6\times10^{-2}$ | | Ebert et al. (2023) | ? | 371 |
| 1-methyl-7-(1-methylethyl)-naphthalene $C_{14}H_{16}$ [490-65-3] UZVVHZJKXPCUQU-UHFFFAOYSA-N | $1.2\times10^{-2}$ | | Parnis et al. (2015) | Q | 369 |
| (*E*)-stilbene $C_{14}H_{12}$ (*trans*-1,2-diphenylethene) [103-30-0] PJANXHGTPQOBST-VAWYXSNFSA-N | $1.4\times10^{-2}$ $1.4\times10^{-2}$ $2.5\times10^{-2}$ $2.5\times10^{-2}$ $2.5\times10^{-2}$ $1.1\times10^{-1}$ | | Duchowicz et al. (2020) HSDB (2015) Mackay et al. (2006a) Mackay et al. (1992b) Duchowicz et al. (2020) Abraham et al. (2019) | V V V X Q Q | 186 364 |
| phenanthrene $C_{14}H_{10}$ [85-01-8] YNPNZTXNASCQKK-UHFFFAOYSA-N | $2.1\times10^{-1}$ $3.4\times10^{-1}$ $2.3\times10^{-1}$ $2.3\times10^{-1}$ $2.3\times10^{-1}$ $2.5\times10^{-1}$ $1.8\times10^{-1}$ $2.7\times10^{-1}$ $2.3\times10^{-1}$ $1.6\times10^{-1}$ $3.4\times10^{-1}$ $2.8\times10^{-1}$ $2.1\times10^{-1}$ $2.5\times10^{-1}$ $4.2\times10^{-1}$ $2.7\times10^{-1}$ $2.5\times10^{-1}$ | 4500 7100 4200 7700 6000 7600 3800 | Schwardt et al. (2021) Brockbank (2013) Ma et al. (2010b) Ma et al. (2010b) Fogg and Sangster (2003) Mackay and Shiu (1981) Lee et al. (2012) Odabasi et al. (2006) Bamford et al. (1999a) Bamford et al. (1999b) De Maagd et al. (1998) Shiu and Mackay (1997) Alaee et al. (1996) Zhang and Pawliszyn (1993) Fendinger and Glotfelty (1990) Mackay and Shiu (1981) Mackay et al. (1979) | L L L L L L M M M M M M M M M M M | 1 1 366 367 12 |



Table A2.7: Polynuclear aromatics (... continued)

| Substance Formula (Trivial Name) [CAS Registry Number] InChIKey | $H_s^{cp}$ (at $T^{\ominus}$) $\left[\dfrac{\mathrm{mol}}{\mathrm{m}^3\,\mathrm{Pa}}\right]$ | $\dfrac{\mathrm{d}\ln H_s^{cp}}{\mathrm{d}(1/T)}$ [K] | Reference | Type | Note |
|---|---|---|---|---|---|
| | $1.8\times10^{-1}$ | | Southworth (1979) | M | |
| | $3.1\times10^{-1}$ | | Mackay et al. (2006a) | V | |
| | $3.1\times10^{-1}$ | | Shiu and Ma (2000) | V | |
| | $3.8\times10^{-1}$ | | De Maagd et al. (1998) | V | 12 |
| | $3.1\times10^{-1}$ | | Shiu and Mackay (1997) | V | |
| | $3.2\times10^{-2}$ | | Hwang et al. (1992) | V | |
| | $2.8\times10^{-1}$ | | Eastcott et al. (1988) | V | |
| | $3.2\times10^{-1}$ | | Cabani et al. (1981) | V | |
| | $2.0\times10^{-1}$ | | Southworth (1979) | V | |
| | $3.9\times10^{-1}$ | | Hine and Mookerjee (1975) | V | |
| | $2.8\times10^{-1}$ | 6500 | Wauchope and Haque (1972) | V | |
| | $9.3\times10^{-2}$ | 4900 | Paasivirta et al. (1999) | T | |
| | $9.0\times10^{-2}$ | | Yaws (2003) | X | 258 |
| | $9.0\times10^{-2}$ | | Yaws (2003) | X | 237 |
| | $9.3\times10^{-2}$ | 4700 | Goldstein (1982) | X | 298 |
| | $7.6\times10^{-2}$ | | McCarty (1980) | X | 368 |
| | $2.5\times10^{-1}$ | | Smith et al. (1993) | C | 294 |
| | $2.5\times10^{-1}$ | | Ryan et al. (1988) | C | |
| | $2.5\times10^{-1}$ | | Dupeux et al. (2022) | Q | 259 |
| | $4.3\times10^{-1}$ | | Keshavarz et al. (2022) | Q | |
| | $9.2\times10^{-2}$ | | Duchowicz et al. (2020) | Q | 184 |
| | $2.1\times10^{-1}$ | | Parnis et al. (2015) | Q | 369 |
| | $7.2\times10^{-2}$ | | Gharagheizi et al. (2010) | Q | 246 |
| | $1.7\times10^{-1}$ | | Schröder et al. (2010) | Q | 363 |
| | $2.6\times10^{-1}$ | | Hilal et al. (2008) | Q | |
| | $6.4\times10^{-1}$ | | Modarresi et al. (2007) | Q | 67 |
| | | 4800 | Kühne et al. (2005) | Q | |
| | $2.5\times10^{-1}$ | | Yaffe et al. (2003) | Q | 248, 249 |
| | $9.2\times10^{-1}$ | | Suzuki et al. (1992) | Q | 232 |
| | $4.8\times10^{-1}$ | | Nirmalakhandan and Speece (1988) | Q | |
| | $2.6\times10^{-1}$ | | Arbuckle (1983) | Q | |
| | $2.3\times10^{-1}$ | | Duchowicz et al. (2020) | ? | 185, 21 |
| | | 5300 | Kühne et al. (2005) | ? | |
| | $2.7\times10^{-1}$ | | Abraham et al. (1990) | ? | |
| 9,10-dihydrophenanthrene $C_{14}H_{12}$ [776-35-2] XXPBFNVKTVJZKF-UHFFFAOYSA-N | $1.2\times10^{-1}$ | 7500 | Reza and Trejo (2004) | M | |
| | $3.2\times10^{-1}$ | | Duchowicz et al. (2020) | V | 186 |
| | $5.8\times10^{-2}$ | | Duchowicz et al. (2020) | Q | |
| | $9.0\times10^{-2}$ | | Schröder et al. (2010) | Q | 363 |
| | $4.1\times10^{-2}$ | | Hilal et al. (2008) | Q | |
| | | 5400 | Kühne et al. (2005) | Q | |
| | | 7500 | Kühne et al. (2005) | ? | |



Table A2.7: Polynuclear aromatics (. . . continued)

| Substance Formula (Trivial Name) [CAS Registry Number] InChIKey | $H_s^{cp}$ (at $T^{\ominus}$) $\left[\dfrac{\text{mol}}{\text{m}^3\,\text{Pa}}\right]$ | $\dfrac{\text{d}\ln H_s^{cp}}{\text{d}(1/T)}$ [K] | Reference | Type | Note |
|---|---|---|---|---|---|
| anthracene | $1.8\times10^{-1}$ | 4800 | Schwardt et al. (2021) | L | 1 |
| $C_{14}H_{10}$ | $3.3\times10^{-1}$ | 6700 | Brockbank (2013) | L | 1 |
| [120-12-7] | $2.0\times10^{-1}$ | | Ma et al. (2010b) | L | 366 |
| MWPLVEDNUUSJAV-UHFFFAOYSA-N | $2.0\times10^{-1}$ | | Ma et al. (2010b) | L | 367 |
| | $1.7\times10^{-1}$ | 5700 | Fogg and Sangster (2003) | L | |
| | $1.7\times10^{-1}$ | | Mackay and Shiu (1981) | L | |
| | $1.6\times10^{-1}$ | | Lee et al. (2012) | M | |
| | $2.3\times10^{-1}$ | 5600 | Reza and Trejo (2004) | M | |
| | $1.8\times10^{-1}$ | 6000 | Bamford et al. (1999a) | M | |
| | $1.5\times10^{-1}$ | 6500 | Bamford et al. (1999b) | M | |
| | $1.3\times10^{-1}$ | | Shiu and Mackay (1997) | M | |
| | $2.0\times10^{-1}$ | 3500 | Alaee et al. (1996) | M | |
| | $1.1\times10^{-1}$ | | Zhang and Pawliszyn (1993) | M | |
| | $5.1\times10^{-1}$ | | Fendinger and Glotfelty (1990) | M | |
| | $2.7\times10^{-1}$ | | Webster et al. (1985) | M | |
| | $1.4\times10^{-2}$ | | Mackay and Shiu (1981) | M | |
| | $1.5\times10^{-1}$ | | Southworth (1979) | M | |
| | $2.5\times10^{-1}$ | | Mackay et al. (2006a) | V | |
| | $2.5\times10^{-1}$ | | Shiu and Ma (2000) | V | |
| | $2.5\times10^{-1}$ | | Shiu and Mackay (1997) | V | |
| | $3.0\times10^{-2}$ | | Hwang et al. (1992) | V | |
| | $6.1\times10^{-1}$ | | Eastcott et al. (1988) | V | |
| | $5.1\times10^{-1}$ | | Cabani et al. (1981) | V | |
| | $3.4\times10^{-2}$ | | Southworth (1979) | V | |
| | $5.6\times10^{-1}$ | | Hine and Mookerjee (1975) | V | |
| | $2.6\times10^{1}$ | 7000 | Wauchope and Haque (1972) | V | |
| | $4.6\times10^{-3}$ | 3100 | Paasivirta et al. (1999) | T | |
| | $3.5\times10^{-1}$ | 4000 | Goldstein (1982) | X | 298 |
| | $7.0\times10^{-3}$ | | McCarty (1980) | X | 368 |
| | $1.1\times10^{-1}$ | | Smith et al. (1993) | C | |
| | $3.7\times10^{-2}$ | | Ryan et al. (1988) | C | |
| | $1.0\times10^{-1}$ | | Smith et al. (1981a) | C | |
| | $4.3\times10^{-1}$ | | Keshavarz et al. (2022) | Q | |
| | $9.2\times10^{-2}$ | | Duchowicz et al. (2020) | Q | |
| | $1.9\times10^{-1}$ | | Parnis et al. (2015) | Q | 369 |
| | $1.5\times10^{-1}$ | | Schröder et al. (2010) | Q | 363 |
| | $3.3\times10^{-1}$ | | Hilal et al. (2008) | Q | |
| | $4.5\times10^{-1}$ | | Modarresi et al. (2007) | Q | 67 |
| | | 6400 | Kühne et al. (2005) | Q | |
| | $1.7\times10^{-1}$ | | Yaffe et al. (2003) | Q | 248, 249 |
| | $7.0\times10^{-1}$ | | Russell et al. (1992) | Q | 279 |
| | $9.0\times10^{-1}$ | | Suzuki et al. (1992) | Q | 232 |
| | $9.0\times10^{-3}$ | | Nirmalakhandan and Speece (1988) | Q | |
| | $1.8\times10^{-1}$ | | Duchowicz et al. (2020) | ? | 185, 21 |
| | | 5100 | Kühne et al. (2005) | ? | |



Table A2.7: Polynuclear aromatics (. . . continued)

| Substance Formula (Trivial Name) [CAS Registry Number] InChIKey | $H_s^{cp}$ (at $T^{\ominus}$) $\left[\dfrac{\text{mol}}{\text{m}^3\,\text{Pa}}\right]$ | $\dfrac{\text{d}\ln H_s^{cp}}{\text{d}(1/T)}$ [K] | Reference | Type | Note |
|---|---|---|---|---|---|
| 9,10-dihydroanthracene $C_{14}H_{12}$ [613-31-0] WPDAVTSOEQEGMS-UHFFFAOYSA-N | $1.5\times10^{-1}$ | | Schröder et al. (2010) | Q | 363 |
| 2,3-dihydro-1,1,3,3,5-pentamethyl-1H-indene | $7.5\times10^{-4}$ | | Zhang et al. (2010) | Q | 287, 288 |
| $C_{14}H_{20}$ | $1.9\times10^{-3}$ | | Zhang et al. (2010) | Q | 287, 289 |
| [81-03-8] | $2.1\times10^{-3}$ | | Zhang et al. (2010) | Q | 287, 290 |
| NNXHDILUOAXSPU-UHFFFAOYSA-N | $3.9\times10^{-4}$ | | Zhang et al. (2010) | Q | 287, 291 |
| 1-methylanthracene $C_{15}H_{12}$ [610-48-0] KZNJSFHJUQDYHE-UHFFFAOYSA-N | $1.7\times10^{-1}$ | | Parnis et al. (2015) | Q | 369 |
| 2-methylanthracene $C_{15}H_{12}$ [613-12-7] GYMFBYTZOGMSQJ-UHFFFAOYSA-N | $1.6\times10^{-2}$ $4.7\times10^{-2}$ | | Duchowicz et al. (2020) Duchowicz et al. (2020) | V Q | 186 |
| 9-methylanthracene $C_{15}H_{12}$ [779-02-2] CPGPAVAKSZHMBP-UHFFFAOYSA-N | $2.5\times10^{-1}$ $6.1\times10^{-1}$ $9.4\times10^{-3}$ $4.7\times10^{-2}$ $1.9\times10^{-1}$ $4.2\times10^{-1}$ | | Duchowicz et al. (2020) Mackay et al. (2006a) Eastcott et al. (1988) Duchowicz et al. (2020) Parnis et al. (2015) Hilal et al. (2008) | V V V Q Q Q | 186 369 |
| 9-methyl-phenanthrene $C_{15}H_{12}$ [883-20-5] DALBHIYZSZZWBS-UHFFFAOYSA-N | $1.9\times10^{-1}$ | | Parnis et al. (2015) | Q | 369 |
| 9-ethylfluorene $C_{15}H_{14}$ [2294-82-8] QBBCCEYJCKGWIK-UHFFFAOYSA-N | $7.4\times10^{-2}$ | | Parnis et al. (2015) | Q | 369 |
| 1,7-dimethylfluorene $C_{15}H_{14}$ [442-66-0] NHPVHXMCRWRSNW-UHFFFAOYSA-N | $9.1\times10^{-2}$ | | Parnis et al. (2015) | Q | 369 |
| 1-methylphenanthrene $C_{15}H_{12}$ [832-69-9] DOWJXOHBNXRUOD-UHFFFAOYSA-N | $2.0\times10^{-1}$ $5.8\times10^{-1}$ $4.7\times10^{-2}$ $1.9\times10^{-1}$ $3.3\times10^{-1}$ $3.8\times10^{-1}$ $2.0\times10^{-1}$ | 4600 5200 | Bamford et al. (1999a) Keshavarz et al. (2022) Duchowicz et al. (2020) Parnis et al. (2015) Hilal et al. (2008) Modarresi et al. (2007) Kühne et al. (2005) Duchowicz et al. (2020) | M Q Q Q Q Q Q ? | 184 369 67 185, 21 |





Table A2.7: Polynuclear aromatics (...continued)

| Substance<br>Formula<br>(Trivial Name)<br>[CAS Registry Number]<br>InChIKey | $H_s^{cp}$<br>(at $T^\ominus$)<br>$\left[\dfrac{\mathrm{mol}}{\mathrm{m^3\,Pa}}\right]$ | $\dfrac{\mathrm{d}\ln H_s^{cp}}{\mathrm{d}(1/T)}$<br><br>[K] | Reference | Type | Note |
|---|---|---|---|---|---|
| | | 4600 | Kühne et al. (2005) | ? | |
| 1,3-diisopropylnaphthalene<br>$C_{16}H_{20}$<br>[57122-16-4]<br>JDBFIFNXALGSOF-UHFFFAOYSA-N | $1.6\times10^{-2}$ | | Schröder et al. (2013) | Q | 370 |
| 1,5-diisopropylnaphthalene<br>$C_{16}H_{20}$<br>[27351-96-8]<br>GFZWCYSEPXDDRI-UHFFFAOYSA-N | $1.6\times10^{-2}$ | | Schröder et al. (2013) | Q | 370 |
| 1,7-diisopropylnaphthalene<br>$C_{16}H_{20}$<br>[94133-80-9]<br>XNSUVOOWWSAQMZ-UHFFFAOYSA-N | $7.8\times10^{-3}$ | | Ebert et al. (2023) | ? | 318 |
| 2,6-diisopropylnaphthalene<br>$C_{16}H_{20}$<br>[24157-81-1]<br>GWLLTEXUIOFAFE-UHFFFAOYSA-N | $1.3\times10^{-2}$ | | Schröder et al. (2013) | Q | 370 |
| 2,7-diisopropylnaphthalene<br>$C_{16}H_{20}$<br>[40458-98-8]<br>YGDMAJYAQCDTNG-UHFFFAOYSA-N | $1.3\times10^{-2}$ | | Schröder et al. (2013) | Q | 370 |
| 2-ethylanthracene<br>$C_{16}H_{14}$<br>[52251-71-5]<br>ZXAGXLDEMUNQSH-UHFFFAOYSA-N | $1.9\times10^{-1}$<br>$4.7\times10^{-2}$ | | Duchowicz et al. (2020)<br>Duchowicz et al. (2020) | V<br>Q | 186 |
| 1,3-dimethylphenanthrene<br>$C_{16}H_{14}$<br>[16664-45-2]<br>UJJUUQKOWULNFNG-UHFFFAOYSA-N | $1.7\times10^{-1}$ | | Parnis et al. (2015) | Q | 369 |
| 3,6-dimethylphenanthrene<br>$C_{16}H_{14}$<br>[1576-67-6]<br>OMIBPZBOAJFEJS-UHFFFAOYSA-N | $1.8\times10^{-1}$ | | Parnis et al. (2015) | Q | 369 |
| 2,3-dimethylanthracene<br>$C_{16}H_{14}$<br>[613-06-9]<br>OGVRJXPGSVLDRD-UHFFFAOYSA-N | $2.1\times10^{-1}$ | | Parnis et al. (2015) | Q | 369 |
| 9,10-dimethylanthracene<br>$C_{16}H_{14}$<br>[781-43-1]<br>JTGMTYWYUZDRBK-UHFFFAOYSA-N | $1.8$<br>$3.4\times10^{-1}$<br>$1.5\times10^{-1}$ | | Mackay et al. (2006a)<br>HSDB (2015)<br>Parnis et al. (2015) | V<br>Q<br>Q | <br>99<br>369 |



Table A2.7: Polynuclear aromatics (...continued)

| Substance<br>Formula<br>(Trivial Name)<br>[CAS Registry Number]<br>InChIKey | $H_s^{cp}$<br>(at $T^\ominus$)<br>$\left[\dfrac{\text{mol}}{\text{m}^3\,\text{Pa}}\right]$ | $\dfrac{\text{d}\ln H_s^{cp}}{\text{d}(1/T)}$<br><br>[K] | Reference | Type | Note |
|---|---|---|---|---|---|
| 9-propyl-9H-fluorene<br>$C_{16}H_{16}$<br>[4037-45-0]<br>ZTBWLZNGULENBG-UHFFFAOYSA-N | $6.1\times10^{-2}$ | | Parnis et al. (2015) | Q | 369 |
| pyrene<br>$C_{16}H_{10}$<br>[129-00-0]<br>BBEAQIROQSPTKN-UHFFFAOYSA-N | $5.9\times10^{-1}$ | 5500 | Schwardt et al. (2021) | L | 1 |
| | 1.1 | 6900 | Brockbank (2013) | L | 1 |
| | $7.5\times10^{-1}$ | | Ma et al. (2010b) | L | 366 |
| | $7.5\times10^{-1}$ | | Ma et al. (2010b) | L | 367 |
| | $6.6\times10^{-1}$ | 4800 | Fogg and Sangster (2003) | L | |
| | $8.3\times10^{-1}$ | | Mackay and Shiu (1981) | L | |
| | $4.1\times10^{-1}$ | | Lee et al. (2012) | M | |
| | $8.5\times10^{-1}$ | 6300 | Reza and Trejo (2004) | M | |
| | 2.0 | | Altschuh et al. (1999) | M | |
| | $5.9\times10^{-1}$ | 5500 | Bamford et al. (1999a) | M | |
| | $5.0\times10^{-1}$ | | De Maagd et al. (1998) | M | 12 |
| | 1.1 | | De Maagd et al. (1998) | M | 12 |
| | $8.3\times10^{-1}$ | | Shiu and Mackay (1997) | M | |
| | $9.1\times10^{-1}$ | | Mackay and Shiu (1981) | M | |
| | $5.3\times10^{-1}$ | | Southworth (1979) | M | |
| | 1.1 | | Mackay et al. (2006a) | V | |
| | 1.1 | | Shiu and Ma (2000) | V | |
| | 1.4 | | De Maagd et al. (1998) | V | 12 |
| | 1.1 | | Shiu and Mackay (1997) | V | |
| | $3.6\times10^{-2}$ | | Hwang et al. (1992) | V | |
| | 1.1 | | Eastcott et al. (1988) | V | |
| | $7.6\times10^{-1}$ | | Cabani et al. (1981) | V | |
| | $9.4\times10^{-1}$ | | Southworth (1979) | V | |
| | 2.2 | 7600 | Wauchope and Haque (1972) | V | |
| | $1.4\times10^{-1}$ | 5700 | Paasivirta et al. (1999) | T | |
| | 1.9 | | Smith et al. (1993) | C | 372 |
| | $1.4\times10^{-3}$ | | Ryan et al. (1988) | C | |
| | 7.6 | | Petrasek et al. (1983) | C | |
| | $7.8\times10^{-1}$ | | Keshavarz et al. (2022) | Q | |
| | $3.0\times10^{-1}$ | | Duchowicz et al. (2020) | Q | 184 |
| | 1.3 | | Abraham et al. (2019) | Q | |
| | $4.7\times10^{-1}$ | | Parnis et al. (2015) | Q | 369 |
| | $3.6\times10^{-1}$ | | Schröder et al. (2010) | Q | 363 |
| | $2.3\times10^{-1}$ | | Hilal et al. (2008) | Q | |
| | 2.3 | | Modarresi et al. (2007) | Q | 67 |
| | | 5200 | Kühne et al. (2005) | Q | |
| | $8.4\times10^{-1}$ | | English and Carroll (2001) | Q | 230, 231 |
| | $5.4\times10^{-1}$ | | Nirmalakhandan and Speece (1988) | Q | |
| | $8.3\times10^{-1}$ | | Duchowicz et al. (2020) | ? | 185, 21 |
| | | 5500 | Kühne et al. (2005) | ? | |
| | $9.0\times10^{-1}$ | | Abraham et al. (1990) | ? | |



Table A2.7: Polynuclear aromatics (...continued)

| Substance Formula (Trivial Name) [CAS Registry Number] InChIKey | $H_s^{cp}$ (at $T^{\ominus}$) $\left[\dfrac{\mathrm{mol}}{\mathrm{m^3\,Pa}}\right]$ | $\dfrac{\mathrm{d\ln}H_s^{cp}}{\mathrm{d}(1/T)}$ [K] | Reference | Type | Note |
|---|---|---|---|---|---|
| benzo[$jk$]fluorene | $5.8\times10^{-1}$ | 5900 | Schwardt et al. (2021) | L | 1 |
| $C_{16}H_{10}$ | $7.0\times10^{-1}$ | 6900 | Brockbank (2013) | L | 1 |
| (fluoranthene) | $6.9\times10^{-1}$ | | Ma et al. (2010b) | L | 366 |
| [206-44-0] | $7.5\times10^{-1}$ | | Ma et al. (2010b) | L | 367 |
| GVEPBJHOBDJJJI-UHFFFAOYSA-N | $5.4\times10^{-1}$ | 4800 | Fogg and Sangster (2003) | L | |
| | $4.5\times10^{-3}$ | | Mackay and Shiu (1981) | L | |
| | $3.4\times10^{-1}$ | | Lee et al. (2012) | M | |
| | $5.1\times10^{-1}$ | 4900 | Bamford et al. (1999a) | M | |
| | $9.1\times10^{-1}$ | | De Maagd et al. (1998) | M | 12 |
| | 1.1 | 6900 | ten Hulscher et al. (1992) | M | |
| | 1.9 | 8700 | Abou-Naccoul et al. (2014) | V | |
| | 1.0 | | Mackay et al. (2006a) | V | |
| | 1.0 | | Shiu and Ma (2000) | V | |
| | 1.4 | | De Maagd et al. (1998) | V | 12 |
| | 1.0 | | Shiu and Mackay (1997) | V | |
| | 2.1 | | McLachlan et al. (1990) | V | 373 |
| | 1.1 | | Eastcott et al. (1988) | V | |
| | $4.0\times10^{-1}$ | 5400 | Paasivirta et al. (1999) | T | |
| | 1.5 | | Smith et al. (1993) | C | |
| | 1.0 | | Ryan et al. (1988) | C | |
| | $9.9\times10^{-1}$ | | Petrasek et al. (1983) | C | |
| | $7.8\times10^{-1}$ | | Keshavarz et al. (2022) | Q | |
| | $3.0\times10^{-1}$ | | Duchowicz et al. (2020) | Q | |
| | 1.1 | | Abraham et al. (2019) | Q | |
| | $5.8\times10^{-1}$ | | Parnis et al. (2015) | Q | 369 |
| | $4.6\times10^{-1}$ | | Schröder et al. (2010) | Q | 363 |
| | $4.4\times10^{-1}$ | | Hilal et al. (2008) | Q | |
| | $2.2\times10^{-1}$ | | Modarresi et al. (2007) | Q | 67 |
| | | 5100 | Kühne et al. (2005) | Q | |
| | 1.1 | | Duchowicz et al. (2020) | ? | 185, 21 |
| | | 5000 | Kühne et al. (2005) | ? | |
| 1,2,3,4-tetrahydro-1,1,3,4,4,6-hexamethylnaphthalene | $4.2\times10^{-4}$ | | Zhang et al. (2010) | Q | 287, 288 |
| $C_{16}H_{24}$ | $1.3\times10^{-3}$ | | Zhang et al. (2010) | Q | 287, 289 |
| [2084-69-7] | $3.2\times10^{-3}$ | | Zhang et al. (2010) | Q | 287, 290 |
| JIVANURAWUCQIG-UHFFFAOYSA-N | $2.7\times10^{-4}$ | | Zhang et al. (2010) | Q | 287, 291 |
| [2.2]paracyclophane | $2.9\times10^{-2}$ | | Zhang et al. (2010) | Q | 287, 288 |
| $C_{16}H_{16}$ | $8.4\times10^{-2}$ | | Zhang et al. (2010) | Q | 287, 289 |
| [1633-22-3] | $9.5\times10^{-1}$ | | Zhang et al. (2010) | Q | 287, 290 |
| OOLUVSIJOMLOCB-UHFFFAOYSA-N | $4.3\times10^{-2}$ | | Zhang et al. (2010) | Q | 287, 291 |





Table A2.7: Polynuclear aromatics (. . . continued)

| Substance Formula (Trivial Name) [CAS Registry Number] InChIKey | $H_s^{cp}$ (at $T^{\ominus}$) $\left[\dfrac{\text{mol}}{\text{m}^3\,\text{Pa}}\right]$ | $\dfrac{\text{d}\ln H_s^{cp}}{\text{d}(1/T)}$ [K] | Reference | Type | Note |
|---|---|---|---|---|---|
| benzo[$a$]fluorene C$_{17}$H$_{12}$ [238-84-6] HKMTVMBEALTRRR-UHFFFAOYSA-N | $3.7\times10^{-1}$ 1.1 $2.2\times10^{-1}$ $4.6\times10^{-1}$ $3.7\times10^{-1}$ | 4400 6300 4400 | Bamford et al. (1999a) Keshavarz et al. (2022) Duchowicz et al. (2020) Modarresi et al. (2007) Kühne et al. (2005) Duchowicz et al. (2020) Kühne et al. (2005) Shiu and Ma (2000) | M Q Q Q Q ? ? W | 184 67 185, 21 360 |
| 1-methylpyrene C$_{17}$H$_{12}$ [2381-21-7] KBSPJIWZDWBDGM-UHFFFAOYSA-N | 3.1 $4.5\times10^{-1}$ | | HSDB (2015) Parnis et al. (2015) | Q Q | 99 369 |
| 2-methylpyrene C$_{17}$H$_{12}$ [3442-78-2] VIRFPLJXRDHVEI-UHFFFAOYSA-N | 3.1 | | HSDB (2015) | Q | 99 |
| 2-methylfluoranthene C$_{17}$H$_{12}$ [30997-39-8] VVRCMNWZFPMXQZ-UHFFFAOYSA-N | $5.4\times10^{-1}$ | | Parnis et al. (2015) | Q | 369 |
| 9-n-propylphenanthrene C$_{17}$H$_{16}$ [17024-03-2] PIWHTUVAMGYSIC-UHFFFAOYSA-N | $1.3\times10^{-1}$ | | Parnis et al. (2015) | Q | 369 |
| 1,2,6-trimethylphenanthrene C$_{17}$H$_{16}$ [30436-55-6] MYWOJODOMFBVCB-UHFFFAOYSA-N | $1.8\times10^{-1}$ | | Parnis et al. (2015) | Q | 369 |
| 9-butyl-9H-fluorene C$_{17}$H$_{18}$ [3952-42-9] RBDADLSAYYPJAN-UHFFFAOYSA-N | $5.1\times10^{-2}$ | | Parnis et al. (2015) | Q | 369 |
| 11H-benzo[$b$]fluorene C$_{17}$H$_{12}$ [243-17-4] HAPOJKSPCGLOOD-UHFFFAOYSA-N | 2.5 $2.2\times10^{-1}$ | | Duchowicz et al. (2020) Duchowicz et al. (2020) | V Q | 186 |
| 2,7-dimethylpyrene C$_{18}$H$_{14}$ [15679-24-0] GSKHIRFMTJUBSM-UHFFFAOYSA-N | 2.9 $3.9\times10^{-1}$ | | HSDB (2015) Parnis et al. (2015) | Q Q | 99 369 |



Table A2.7: Polynuclear aromatics (. . . continued)

| Substance Formula (Trivial Name) [CAS Registry Number] InChIKey | $H_s^{cp}$ (at $T^{\ominus}$) $\left[\dfrac{\mathrm{mol}}{\mathrm{m^3\,Pa}}\right]$ | $\dfrac{\mathrm{d}\ln H_s^{cp}}{\mathrm{d}(1/T)}$ [K] | Reference | Type | Note |
|---|---|---|---|---|---|
| chrysene | $1.5\times10^1$ | 12000 | Brockbank (2013) | L | 1 |
| $C_{18}H_{12}$ | 2.3 | | Ma et al. (2010b) | L | 366 |
| [218-01-9] | 2.7 | | Ma et al. (2010b) | L | 367 |
| WDECIBYCCFPHNR-UHFFFAOYSA-N | 2.1 | | Lee et al. (2012) | M | |
| | 1.9 | 13000 | Bamford et al. (1999a) | M | |
| | 9.4 | | Zhang and Pawliszyn (1993) | M | |
| | $1.0\times10^1$ | | HSDB (2015) | V | |
| | $1.5\times10^1$ | | Mackay et al. (2006a) | V | |
| | $1.5\times10^1$ | | Shiu and Ma (2000) | V | |
| | 2.2 | | Eastcott et al. (1988) | V | |
| | $2.0\times10^{-1}$ | 6400 | Paasivirta et al. (1999) | T | |
| | 9.4 | | Smith et al. (1993) | C | |
| | $4.6\times10^{-3}$ | | Ryan et al. (1988) | C | |
| | 6.6 | | Petrasek et al. (1983) | C | |
| | 1.4 | | Keshavarz et al. (2022) | Q | |
| | $3.6\times10^{-1}$ | | Duchowicz et al. (2020) | Q | 299 |
| | 2.4 | | Parnis et al. (2015) | Q | 369 |
| | 2.0 | | Schröder et al. (2010) | Q | 363 |
| | 3.6 | | Hilal et al. (2008) | Q | |
| | 5.0 | | Modarresi et al. (2007) | Q | 67 |
| | 1.9 | | Duchowicz et al. (2020) | ? | 185, 21 |
| naphthacene | $2.0\times10^1$ | | Duchowicz et al. (2020) | V | 186 |
| $C_{18}H_{12}$ | $3.6\times10^2$ | | Mackay et al. (2006a) | V | |
| (2,3-benzanthracene) | $2.5\times10^2$ | | Mackay et al. (1992b) | X | 364 |
| [92-24-0] | $3.6\times10^{-1}$ | | Duchowicz et al. (2020) | Q | |
| IFLREYGFSNHWGE-UHFFFAOYSA-N | 4.2 | | Ferreira (2001) | Q | 12 |
| triphenylene | $6.4\times10^1$ | | Duchowicz et al. (2020) | V | 186 |
| $C_{18}H_{12}$ | | | Mackay et al. (2006a) | V | 292 |
| (benzo[$l$]phenanthrene) | $1.0\times10^2$ | | Mackay et al. (1992b) | X | 364 |
| [217-59-4] | $3.6\times10^{-1}$ | | Duchowicz et al. (2020) | Q | |
| SLGBZMMZGDRARJ-UHFFFAOYSA-N | 8.6 | | Schröder et al. (2010) | Q | 363 |
| | 2.9 | | Hilal et al. (2008) | Q | |
| | 4.7 | | Modarresi et al. (2007) | Q | 67 |
| | 3.1 | | Ferreira (2001) | Q | 12 |
| benz[$a$]anthracene | 1.4 | | Ma et al. (2010b) | L | 366 |
| $C_{18}H_{12}$ | 1.6 | | Ma et al. (2010b) | L | 367 |
| [56-55-3] | $9.0\times10^{-1}$ | 7900 | Fogg and Sangster (2003) | L | |
| DXBHBZVCASKNBY-UHFFFAOYSA-N | 1.7 | | Lee et al. (2012) | M | |
| | $8.2\times10^{-1}$ | 8300 | Bamford et al. (1999a) | M | |
| | 9.9 | | Zhang and Pawliszyn (1993) | M | |
| | 1.2 | | Southworth (1979) | M | |
| | 1.7 | | Mackay et al. (2006a) | V | |
| | 2.4 | | Eastcott et al. (1988) | V | |
| | $7.5\times10^1$ | | Smith and Bomberger (1980) | V | 24 |
| | 4.0 | | Southworth (1979) | V | |
| | $1.5\times10^{-1}$ | 6100 | Paasivirta et al. (1999) | T | |



Table A2.7: Polynuclear aromatics (...continued)

| Substance Formula (Trivial Name) [CAS Registry Number] InChIKey | $H_s^{cp}$ (at $T^{\ominus}$) $\left[\dfrac{\text{mol}}{\text{m}^3\,\text{Pa}}\right]$ | $\dfrac{\text{d}\ln H_s^{cp}}{\text{d}(1/T)}$ [K] | Reference | Type | Note |
|---|---|---|---|---|---|
| | 8.5 | | Smith et al. (1993) | C | 79 |
| | 9.8 | | Ryan et al. (1988) | C | |
| | $8.2\times10^1$ | | Petrasek et al. (1983) | C | |
| | 1.4 | | Keshavarz et al. (2022) | Q | |
| | $3.6\times10^{-1}$ | | Duchowicz et al. (2020) | Q | |
| | 2.2 | | Parnis et al. (2015) | Q | 369 |
| | 1.6 | | Schröder et al. (2010) | Q | 363 |
| | 4.4 | | Hilal et al. (2008) | Q | |
| | 5.0 | | Modarresi et al. (2007) | Q | 67 |
| | | 6100 | Kühne et al. (2005) | Q | |
| | 5.6 | | Ferreira (2001) | Q | 12 |
| | $8.2\times10^{-1}$ | | Duchowicz et al. (2020) | ? | 185, 21 |
| | | 8300 | Kühne et al. (2005) | ? | |
| | | | Shiu and Ma (2000) | W | 360 |
| 4,5-dimethylpyrene $C_{18}H_{14}$ [15679-25-1] MCZKPUHBCIHFBZ-UHFFFAOYSA-N | $4.5\times10^{-1}$ | | Parnis et al. (2015) | Q | 369 |
| 3-ethylfluoranthene $C_{18}H_{14}$ [20496-16-6] JXUOLJXLPIIRTP-UHFFFAOYSA-N | $4.8\times10^{-1}$ | | Parnis et al. (2015) | Q | 369 |
| 1,2,6,9-tetramethylphenanthrene $C_{18}H_{18}$ [204256-39-3] XFJWRZAPBOJONX-UHFFFAOYSA-N | $1.6\times10^{-1}$ | | Parnis et al. (2015) | Q | 369 |
| 1,9-dimethyl-5-ethylphenanthrene $C_{18}H_{18}$ MDZYWNWRMJIOPN-UHFFFAOYSA-N | $1.0\times10^{-1}$ | | Parnis et al. (2015) | Q | 369 |
| 1,9-dimethyl-7-ethylphenanthrene $C_{18}H_{18}$ NFFCDFHAMXFVEK-UHFFFAOYSA-N | $1.4\times10^{-1}$ | | Parnis et al. (2015) | Q | 369 |
| 1,2,3,4-tetrahydro-5-(1-phenylethyl)-naphthalene $C_{18}H_{20}$ [60466-61-7] TXOHWLOHKUPUKO-UHFFFAOYSA-N | $1.6\times10^{-2}$ $1.0\times10^{-1}$ $2.0\times10^{-1}$ $2.9\times10^{-2}$ | | Zhang et al. (2010) Zhang et al. (2010) Zhang et al. (2010) Zhang et al. (2010) | Q Q Q Q | 287, 288 287, 289 287, 290 287, 291 |
| 4-methylchrysene $C_{19}H_{14}$ [3351-30-2] BLVHWJCLSMYFMT-UHFFFAOYSA-N | 1.9 | | Parnis et al. (2015) | Q | 369 |



Table A2.7: Polynuclear aromatics (. . . continued)

| Substance Formula (Trivial Name) [CAS Registry Number] InChIKey | $H_s^{cp}$ (at $T^\ominus$) $\left[\dfrac{\text{mol}}{\text{m}^3\,\text{Pa}}\right]$ | $\dfrac{\text{d}\ln H_s^{cp}}{\text{d}(1/T)}$ [K] | Reference | Type | Note |
|---|---|---|---|---|---|
| 5-methylchrysene $C_{19}H_{14}$ [3697-24-3] GOHBXWHNJHENRX-UHFFFAOYSA-N | 5.2 | | HSDB (2015) | Q | 99 |
| 6-methylchrysene $C_{19}H_{14}$ [1705-85-7] ASVDRLYVNFOSCI-UHFFFAOYSA-N | 2.2 | | Parnis et al. (2015) | Q | 369 |
| 1-propylpyrene $C_{19}H_{16}$ [42211-33-6] HIOZPFGGCXEPAP-UHFFFAOYSA-N | $3.2\times10^{-1}$ | | Parnis et al. (2015) | Q | 369 |
| 7-methylbenz[$a$]anthracene $C_{19}H_{14}$ [2541-69-7] DIIFUCUPDHMNIV-UHFFFAOYSA-N | 5.2 | | HSDB (2015) | Q | 99 |
| 10-methylbenz[$a$]anthracene $C_{19}H_{14}$ [2381-15-9] WUMGYHICFXGLAB-UHFFFAOYSA-N | 5.2 | | HSDB (2015) | Q | 99 |
| 12-methylbenz[$a$]anthracene $C_{19}H_{14}$ [2422-79-9] ACYOLKMEHHTLAB-UHFFFAOYSA-N | 5.2 | | HSDB (2015) | Q | 99 |
| 6-ethylchrysene $C_{20}H_{16}$ [2732-58-3] ZJSYTTGSPQNXKT-UHFFFAOYSA-N | 1.8 | | Parnis et al. (2015) | Q | 369 |
| 3,9-dimethylbenz[$a$]anthracene $C_{20}H_{16}$ [316-51-8] DBPDWIZRMPUWRH-UHFFFAOYSA-N | 1.9 | | Parnis et al. (2015) | Q | 369 |
| 1-butylpyrene $C_{20}H_{18}$ [35980-18-8] UFOTZLIYHMGVAV-UHFFFAOYSA-N | $2.8\times10^{-1}$ | | Parnis et al. (2015) | Q | 369 |
| 7,12-dimethyl-benz[$a$]anthracene $C_{20}H_{16}$ [57-97-6] ARSRBNBHOADGJU-UHFFFAOYSA-N | 2.6 $5.1\times10^{3}$ $9.2\times10^{-2}$ 4.9 1.5 | | Duchowicz et al. (2020) Mackay et al. (2006a) Duchowicz et al. (2020) HSDB (2015) Parnis et al. (2015) | V V Q Q Q | 186 99 369 |



Table A2.7: Polynuclear aromatics (...continued)

| Substance Formula (Trivial Name) [CAS Registry Number] InChIKey | $H_s^{cp}$ (at $T^{\ominus}$) $\left[\dfrac{\mathrm{mol}}{\mathrm{m^3\,Pa}}\right]$ | $\dfrac{\mathrm{d}\ln H_s^{cp}}{\mathrm{d}(1/T)}$ [K] | Reference | Type | Note |
|---|---|---|---|---|---|
| 9,10-dimethyl-benz[a]anthracene $C_{20}H_{16}$ [58429-99-5] GKVUDAZMZLMNJQ-UHFFFAOYSA-N | | | Mackay et al. (2006a) | V | 292 |
| benzo[b]fluoranthene $C_{20}H_{12}$ [205-99-2] FTOVXSOBNPWTSH-UHFFFAOYSA-N | $1.5\times10^1$ | 5400 | Brockbank (2013) | L | 1 |
| | $1.5\times10^1$ | | Ma et al. (2010b) | L | 366 |
| | $1.5\times10^1$ | | Ma et al. (2010b) | L | 367 |
| | $1.5\times10^1$ | 5400 | ten Hulscher et al. (1992) | M | |
| | $1.4\times10^1$ | 7500 | Paasivirta et al. (1999) | T | |
| | $8.3\times10^{-1}$ | | Smith et al. (1993) | C | |
| | 1.4 | | Keshavarz et al. (2022) | Q | |
| | 1.2 | | Duchowicz et al. (2020) | Q | 184 |
| | 6.0 | | Parnis et al. (2015) | Q | 369 |
| | 5.6 | | Hilal et al. (2008) | Q | |
| | 2.1 | | Modarresi et al. (2007) | Q | 67 |
| | | 4700 | Kühne et al. (2005) | Q | |
| | $2.0\times10^{-2}$ | | Yaffe et al. (2003) | Q | 248, 249 |
| | $1.5\times10^1$ | | Duchowicz et al. (2020) | ? | 185, 21 |
| | | 5400 | Kühne et al. (2005) | ? | |
| benzo[k]fluoranthene $C_{20}H_{12}$ [207-08-9] HAXBIWFMXWRORI-UHFFFAOYSA-N | $1.7\times10^1$ | | Ma et al. (2010b) | L | 366 |
| | $1.8\times10^1$ | | Ma et al. (2010b) | L | 367 |
| | $1.0\times10^1$ | | Lee et al. (2012) | M | |
| | $1.7\times10^1$ | 5900 | ten Hulscher et al. (1992) | M | |
| | | | Mackay et al. (2006a) | V | 292 |
| | $8.3\times10^1$ | | De Maagd et al. (1998) | V | 12 |
| | $6.2\times10^1$ | | Shiu and Mackay (1997) | V | |
| | 1.5 | 6900 | Paasivirta et al. (1999) | T | |
| | $9.6\times10^{-3}$ | 1900 | Goldstein (1982) | X | 298 |
| | $2.5\times10^{-1}$ | | Smith et al. (1993) | C | |
| | 2.6 | | Keshavarz et al. (2022) | Q | |
| | 1.2 | | Duchowicz et al. (2020) | Q | 299 |
| | 5.9 | | Parnis et al. (2015) | Q | 369 |
| | 8.0 | | Hilal et al. (2008) | Q | |
| | 2.4 | | Modarresi et al. (2007) | Q | 67 |
| | | 6300 | Kühne et al. (2005) | Q | |
| | $2.0\times10^{-2}$ | | Yaffe et al. (2003) | Q | 248, 272 |
| | $1.7\times10^1$ | | Duchowicz et al. (2020) | ? | 185, 21 |
| | | 5800 | Kühne et al. (2005) | ? | |
| benzo[a]pyrene $C_{20}H_{12}$ (benz[a]pyrene) [50-32-8] FMMWHPNWAFZXNH-UHFFFAOYSA-N | 4.5 | 8500 | Brockbank (2013) | L | 1 |
| | $2.0\times10^1$ | | Ma et al. (2010b) | L | 366 |
| | $1.3\times10^1$ | | Ma et al. (2010b) | L | 367 |
| | 6.2 | | Lee et al. (2012) | M | |
| | $1.3\times10^1$ | | Altschuh et al. (1999) | M | |
| | $2.2\times10^1$ | 4700 | ten Hulscher et al. (1992) | M | |
| | $2.2\times10^1$ | | Mackay et al. (2006a) | V | |
| | $2.9\times10^1$ | | De Maagd et al. (1998) | V | 12 |




Table A2.7: Polynuclear aromatics (...continued)

| Substance Formula (Trivial Name) [CAS Registry Number] InChIKey | $H_s^{cp}$ (at $T^\ominus$) $\left[\dfrac{\mathrm{mol}}{\mathrm{m^3\,Pa}}\right]$ | $\dfrac{\mathrm{d}\ln H_s^{cp}}{\mathrm{d}(1/T)}$ [K] | Reference | Type | Note |
|---|---|---|---|---|---|
| | $2.2\times10^1$ | | Shiu and Mackay (1997) | V | |
| | $1.3\times10^2$ | | McLachlan et al. (1990) | V | 373 |
| | $1.8\times10^1$ | | Eastcott et al. (1988) | V | |
| | $7.5$ | | Smith and Bomberger (1980) | V | 24 |
| | $1.9\times10^1$ | | Southworth (1979) | V | |
| | $8.2\times10^{-1}$ | 8200 | Paasivirta et al. (1999) | T | |
| | $1.6\times10^{-3}$ | 110 | Goldstein (1982) | X | 298 |
| | $2.0\times10^1$ | | Smith et al. (1993) | C | |
| | $8.2\times10^{-4}$ | | Ryan et al. (1988) | C | |
| | $2.6$ | | Keshavarz et al. (2022) | Q | |
| | $1.2$ | | Duchowicz et al. (2020) | Q | 299 |
| | $5.1$ | | Parnis et al. (2015) | Q | 369 |
| | $2.9$ | | Hilal et al. (2008) | Q | |
| | $1.5\times10^1$ | | Modarresi et al. (2007) | Q | 67 |
| | | 4900 | Kühne et al. (2005) | Q | |
| | $2.9\times10^{-2}$ | | Yaffe et al. (2003) | Q | 248, 249 |
| | $2.2\times10^1$ | | Duchowicz et al. (2020) | ? | 185, 21 |
| | | 4700 | Kühne et al. (2005) | ? | |
| | | | Shiu and Ma (2000) | W | 360 |
| benzo[$e$]pyrene C$_{20}$H$_{12}$ [192-97-2] TXVHTIQJNYSSKO-UHFFFAOYSA-N | $3.3\times10^1$ | | Duchowicz et al. (2020) | V | 186 |
| | $3.3\times10^1$ | | HSDB (2015) | V | |
| | $2.1\times10^1$ | | Mackay et al. (2006a) | V | |
| | $2.7$ | 8300 | Paasivirta et al. (1999) | T | |
| | $1.2$ | | Duchowicz et al. (2020) | Q | |
| | $1.5\times10^1$ | | Ferreira (2001) | Q | 12 |
| | | | Shiu and Ma (2000) | W | 360 |
| perylene C$_{20}$H$_{12}$ (dibenz[$de,kl$]anthracene) [198-55-0] CSHWQDPOILHKBI-UHFFFAOYSA-N | $2.7$ | | Duchowicz et al. (2020) | V | 186 |
| | | | Mackay et al. (2006a) | V | 292 |
| | $2.3$ | | Riederer (1990) | V | |
| | $2.5\times10^{-1}$ | 6300 | Paasivirta et al. (1999) | T | |
| | $3.3\times10^2$ | | Mackay et al. (1992b) | X | 364 |
| | $1.2$ | | Duchowicz et al. (2020) | Q | |
| | $5.7$ | | Parnis et al. (2015) | Q | 369 |
| | $2.3$ | | Hilal et al. (2008) | Q | |
| | $1.1\times10^1$ | | Ferreira (2001) | Q | 12 |
| 1,2-benzfluoranthene C$_{20}$H$_{12}$ [203-33-8] OQDXASJSCOTNQS-UHFFFAOYSA-N | $6.9$ | | Hilal et al. (2008) | Q | |
| benzo[$j$]fluoranthene C$_{20}$H$_{12}$ [205-82-3] KHNYNFUTFKJLDD-UHFFFAOYSA-N | $4.9\times10^1$ | | HSDB (2015) | Q | 99 |





Table A2.7: Polynuclear aromatics (...continued)

| Substance Formula (Trivial Name) [CAS Registry Number] InChIKey | $H_s^{cp}$ (at $T^\ominus$) $\left[\dfrac{\text{mol}}{\text{m}^3\,\text{Pa}}\right]$ | $\dfrac{\text{d}\ln H_s^{cp}}{\text{d}(1/T)}$ [K] | Reference | Type | Note |
|---|---|---|---|---|---|
| 1,3,6-trimethylchrysene $C_{21}H_{18}$ [1586755-28-3] UGLPREDFUVNEMJ-UHFFFAOYSA-N | 1.8 | | Parnis et al. (2015) | Q | 369 |
| 6-propylchrysene $C_{21}H_{18}$ [6910-41-4] PGBSMHMYMMBHFF-UHFFFAOYSA-N | 1.5 | | Parnis et al. (2015) | Q | 369 |
| 20-methylcholanthrene $C_{21}H_{16}$ [56-49-5] PPQNQXQZIWHJRB-UHFFFAOYSA-N | 1.9 1.9 $3.7\times10^{-1}$ | | Duchowicz et al. (2020) HSDB (2015) Mackay et al. (2006a) Duchowicz et al. (2020) | V V V Q | 186 292 |
| 6-butylchrysene $C_{22}H_{20}$ [6901-71-9] XJQKMLMKECCWEO-UHFFFAOYSA-N | 1.3 | | Parnis et al. (2015) | Q | 369 |
| dibenz[$a,h$]anthracene $C_{22}H_{14}$ [53-70-3] LHRCREOYAASXPZ-UHFFFAOYSA-N | $7.0\times10^1$ $1.8\times10^2$ $5.8\times10^3$ $1.3\times10^2$ 1.2 $1.4\times10^2$ 1.4 $1.4\times10^2$ $2.6\times10^1$ $1.2\times10^1$ $8.3\times10^1$ | 12000 7800 | Duchowicz et al. (2020) Abou-Naccoul et al. (2014) Mackay et al. (2006a) Eastcott et al. (1988) Paasivirta et al. (1999) Smith et al. (1993) Duchowicz et al. (2020) HSDB (2015) Parnis et al. (2015) Hilal et al. (2008) Ferreira (2001) | V V V V T C Q Q Q Q Q | 186 99 369 12 |
| indeno[1,2,3-$cd$]pyrene $C_{22}H_{12}$ [193-39-5] SXQBHARYMNFBPS-UHFFFAOYSA-N | $2.9\times10^1$ $2.0\times10^1$ $2.8\times10^1$ 2.5 $1.4\times10^2$ 4.7 3.7 $1.3\times10^1$ 5.0 9.4 $2.8\times10^1$ | 3600 7400 5100 5100 3600 | Ma et al. (2010b) Ma et al. (2010b) ten Hulscher et al. (1992) Paasivirta et al. (1999) Smith et al. (1993) Keshavarz et al. (2022) Duchowicz et al. (2020) Parnis et al. (2015) Hilal et al. (2008) Modarresi et al. (2007) Kühne et al. (2005) Duchowicz et al. (2020) Kühne et al. (2005) | L L M T C Q Q Q Q Q Q ? ? | 366 367 184 369 67 185, 21 |





Table A2.7: Polynuclear aromatics (…continued)

| Substance Formula (Trivial Name) [CAS Registry Number] InChIKey | $H_s^{cp}$ (at $T^{\ominus}$) $\left[\dfrac{\mathrm{mol}}{\mathrm{m^3\,Pa}}\right]$ | $\dfrac{\mathrm{d}\ln H_s^{cp}}{\mathrm{d}(1/T)}$ [K] | Reference | Type | Note |
|---|---|---|---|---|---|
| benzo[$ghi$]perylene | $3.0\times10^1$ | | Ma et al. (2010b) | L | 366 |
| C$_{22}$H$_{12}$ | $2.4\times10^1$ | | Ma et al. (2010b) | L | 367 |
| [191-24-2] | $3.0\times10^1$ | 3200 | ten Hulscher et al. (1992) | M | |
| GYFAGKUZYNFMBN-UHFFFAOYSA-N | $1.8\times10^1$ | | De Maagd et al. (1998) | V | 12 |
| | $1.3\times10^1$ | | Shiu and Mackay (1997) | V | |
| | $6.9\times10^1$ | | Eastcott et al. (1988) | V | |
| | 4.0 | 9200 | Paasivirta et al. (1999) | T | |
| | $1.3\times10^1$ | | Mackay et al. (1992b) | X | 364 |
| | $1.8\times10^2$ | | Smith et al. (1993) | C | |
| | 4.7 | | Keshavarz et al. (2022) | Q | |
| | 3.7 | | Duchowicz et al. (2020) | Q | |
| | $1.1\times10^1$ | | Parnis et al. (2015) | Q | 369 |
| | 2.6 | | Hilal et al. (2008) | Q | |
| | $6.1\times10^1$ | | Modarresi et al. (2007) | Q | 67 |
| | | 3700 | Kühne et al. (2005) | Q | |
| | $3.7\times10^{-2}$ | | Yaffe et al. (2003) | Q | 248, 249 |
| | $3.0\times10^1$ | | Duchowicz et al. (2020) | ? | 185, 21 |
| | | 3300 | Kühne et al. (2005) | ? | |
| benzo[$b$]triphenylene | $1.9\times10^1$ | 8600 | Abou-Naccoul et al. (2014) | V | |
| C$_{22}$H$_{14}$ | $4.4\times10^3$ | | Mackay et al. (2006a) | V | |
| (dibenz[$a,c$]anthracene) | $1.9\times10^1$ | | Hilal et al. (2008) | Q | |
| [215-58-7] | $1.4\times10^2$ | | Ferreira (2001) | Q | 12 |
| RAASUWZPTOJQAY-UHFFFAOYSA-N | | | | | |
| dibenz[$a,j$]anthracene | $8.6\times10^1$ | | Hilal et al. (2008) | Q | |
| C$_{22}$H$_{14}$ | $8.3\times10^1$ | | Ferreira (2001) | Q | 12 |
| [224-41-9] | | | | | |
| KLIHYVJAYWCEDM-UHFFFAOYSA-N | | | | | |
| picene | 6.2 | | Hilal et al. (2008) | Q | |
| C$_{22}$H$_{14}$ | $7.7\times10^1$ | | Ferreira (2001) | Q | 12 |
| [213-46-7] | | | | | |
| GBROPGWFBFCKAG-UHFFFAOYSA-N | | | | | |
| benzo[$c$]chrysene | $8.0\times10^1$ | | HSDB (2015) | Q | 99 |
| C$_{22}$H$_{14}$ | | | | | |
| [194-69-4] | | | | | |
| YZWGEMSQAMDWEM-UHFFFAOYSA-N | | | | | |
| benzo[$g$]chrysene | $8.0\times10^1$ | | HSDB (2015) | Q | 99 |
| C$_{22}$H$_{14}$ | | | | | |
| [196-78-1] | | | | | |
| JZOIZKBKSZMVRV-UHFFFAOYSA-N | | | | | |
| dibenzo[$a,e$]pyrene | $7.0\times10^2$ | | HSDB (2015) | Q | 99 |
| C$_{24}$H$_{14}$ | | | | | |
| [192-65-4] | | | | | |
| KGHMWBNEMFNJFZ-UHFFFAOYSA-N | | | | | |



Table A2.7: Polynuclear aromatics (...continued)

| Substance Formula (Trivial Name) [CAS Registry Number] InChIKey | $H_s^{cp}$ (at $T^{\ominus}$) $\left[\dfrac{\mathrm{mol}}{\mathrm{m}^3\,\mathrm{Pa}}\right]$ | $\dfrac{\mathrm{d}\ln H_s^{cp}}{\mathrm{d}(1/T)}$ [K] | Reference | Type | Note |
|---|---|---|---|---|---|
| dibenzo[$a,h$]pyrene C$_{24}$H$_{14}$ [189-64-0] RXUSYFJGDZFVND-UHFFFAOYSA-N | $7.0{\times}10^2$ | | HSDB (2015) | Q | 99 |
| dibenzo[$a,i$]pyrene C$_{24}$H$_{14}$ [189-55-9] TUGYIJVAYAHHHM-UHFFFAOYSA-N | $7.0{\times}10^2$ | | HSDB (2015) | Q | 99 |
| coronene C$_{24}$H$_{12}$ [191-07-1] VPUGDVKSAQVFFS-UHFFFAOYSA-N | | | Mackay et al. (2006a) | V | 292 |
| dibenz[$a,e$]aceanthrylene C$_{24}$H$_{14}$ [5385-75-1] JHOWUOKQHJHGMU-UHFFFAOYSA-N | $7.0{\times}10^2$ | | HSDB (2015) | Q | 99 |
| dibenzo[$b,k$]chrysene C$_{26}$H$_{16}$ [217-54-9] DHCSBRKYHMINPB-UHFFFAOYSA-N | $1.2{\times}10^3$ | | HSDB (2015) | Q | 99 |



## A3   Organic species with oxygen (O)

### A3.1   Carbon oxides

Table A3.1: Carbon oxides

| Substance Formula (Trivial Name) [CAS Registry Number] InChIKey | $H_s^{cp}$ (at $T^\ominus$) $\left[\dfrac{\text{mol}}{\text{m}^3\,\text{Pa}}\right]$ | $\dfrac{\text{d}\ln H_s^{cp}}{\text{d}(1/T)}$ [K] | Reference | Type | Note |
|---|---|---|---|---|---|
| carbon monoxide | $9.7\times10^{-6}$ | 1300 | Burkholder et al. (2019) | L | 1 |
| CO | $8.0\times10^{-6}$ | 1400 | Burkholder et al. (2019) | L | 70 |
| [630-08-0] | $9.7\times10^{-6}$ | 1300 | Burkholder et al. (2015) | L | 1 |
| UGFAIRIUMAVXCW-UHFFFAOYSA-N | $8.0\times10^{-6}$ | 1400 | Burkholder et al. (2015) | L | 70 |
| | $9.7\times10^{-6}$ | 1300 | Warneck and Williams (2012) | L | |
| | $9.7\times10^{-6}$ | 1300 | Sander et al. (2011) | L | 1 |
| | $9.7\times10^{-6}$ | 1300 | Sander et al. (2006) | L | 1 |
| | $9.7\times10^{-6}$ | 1300 | Fernández-Prini et al. (2003) | L | 3 |
| | | | Cargill (1990) | L | 374 |
| | $9.4\times10^{-6}$ | 1300 | Wilhelm et al. (1977) | L | |
| | $9.7\times10^{-6}$ | 1300 | Rettich et al. (1982) | M | 375 |
| | $7.0\times10^{-6}$ | 1500 | Schmidt (1979) | M | 33, 34 |
| | $7.9\times10^{-5}$ | | Meadows and Spedding (1974) | M | |
| | $8.3\times10^{-6}$ | | Power and Stegall (1970) | M | 14 |
| | $7.9\times10^{-6}$ | 1200 | Douglas (1967) | M | 376, 377 |
| | $9.4\times10^{-6}$ | 1300 | Winkler (1901) | M | 378 |
| | $9.5\times10^{-6}$ | 1200 | Bunsen (1855a) | M | 43 |
| | $6.0\times10^{-6}$ | | Pierotti (1965) | T | |
| | $1.0\times10^{-5}$ | | Hayer et al. (2022) | Q | 20 |
| | $8.7\times10^{-6}$ | | Yaws (1999) | ? | 21 |
| | $9.4\times10^{-6}$ | 1300 | Yaws et al. (1999) | ? | 21 |
| | $8.4\times10^{-6}$ | | Abraham and Weathersby (1994) | ? | 21 |
| | $9.4\times10^{-6}$ | 1400 | Dean and Lange (1999) | ? | 379, 23 |
| | $8.6\times10^{-6}$ | | Yaws and Yang (1992) | ? | 21 |
| carbon dioxide | $3.4\times10^{-4}$ | 2300 | Burkholder et al. (2019) | L | 1 |
| $CO_2$ | $2.8\times10^{-4}$ | 2600 | Burkholder et al. (2019) | L | 70 |
| [124-38-9] | $3.4\times10^{-4}$ | 2300 | Burkholder et al. (2015) | L | 1 |
| CURLTUGMZLYLDI-UHFFFAOYSA-N | $2.8\times10^{-4}$ | 2600 | Burkholder et al. (2015) | L | 70 |
| | $3.3\times10^{-4}$ | 2400 | Sander et al. (2011) | L | 1 |
| | $3.3\times10^{-4}$ | 2400 | Sander et al. (2006) | L | 1 |
| | $3.3\times10^{-4}$ | 2300 | Fernández-Prini et al. (2003) | L | 3 |
| | $3.4\times10^{-4}$ | 2300 | Carroll et al. (1991) | L | |
| | $3.4\times10^{-4}$ | 2400 | Crovetto (1991) | L | |
| | $3.4\times10^{-4}$ | 2300 | Yoo et al. (1986) | L | 1 |
| | $3.4\times10^{-4}$ | 2400 | Edwards et al. (1978) | L | 1 |
| | $3.3\times10^{-4}$ | 2400 | Wilhelm et al. (1977) | L | |
| | $3.4\times10^{-4}$ | 2400 | Weiss (1974) | L | 1 |
| | $3.4\times10^{-4}$ | 2300 | Zheng et al. (1997) | M | 380 |
| | $3.3\times10^{-4}$ | 2400 | Murray and Riley (1971) | M | 381 |
| | $2.4\times10^{-4}$ | | Power and Stegall (1970) | M | 14 |
| | $3.3\times10^{-4}$ | 2400 | Morrison and Billett (1952) | M | 382 |
| | $3.3\times10^{-4}$ | | Orcutt and Seevers (1937a) | M | |



Table A3.1: Carbon oxides (...continued)

| Substance / Formula / (Trivial Name) / [CAS Registry Number] / InChIKey | $H_s^{cp}$ (at $T^{\ominus}$) $\left[\dfrac{\text{mol}}{\text{m}^3\,\text{Pa}}\right]$ | $\dfrac{\text{d}\ln H_s^{cp}}{\text{d}(1/T)}$ [K] | Reference | Type | Note |
|---|---|---|---|---|---|
| | $3.3\times10^{-4}$ | 2300 | Kunerth (1922) | M | |
| | $3.3\times10^{-4}$ | 2500 | Geffcken (1904) | M | |
| | $3.4\times10^{-4}$ | 2400 | Bohr (1899) | M | 383 |
| | $3.4\times10^{-4}$ | 2500 | Bunsen (1855a) | M | 43 |
| | $6.5\times10^{-4}$ | | Duchowicz et al. (2020) | V | 186 |
| | $3.4\times10^{-4}$ | 2400 | Chen et al. (1979) | R | 1 |
| | $3.1\times10^{-4}$ | 2400 | Chameides (1984) | T | |
| | $3.3\times10^{-4}$ | 2400 | Edwards et al. (1975) | T | 1 |
| | $3.4\times10^{-4}$ | | Perry and Chilton (1973) | X | 29 |
| | $3.4\times10^{-4}$ | 2400 | Lelieveld and Crutzen (1991) | C | |
| | $3.4\times10^{-4}$ | 2400 | Pandis and Seinfeld (1989) | C | |
| | $3.9\times10^{-4}$ | | Nunn (1958) | C | 12 |
| | $2.3\times10^{-4}$ | | Hayer et al. (2022) | Q | 20 |
| | 4.0 | | Duchowicz et al. (2020) | Q | |
| | | 2900 | Kühne et al. (2005) | Q | |
| | | | Scharlin (1996) | E | 1, 384 |
| | | 2400 | Kühne et al. (2005) | ? | |
| | $4.5\times10^{-4}$ | | Yaws (1999) | ? | 21 |
| | $3.3\times10^{-4}$ | 2400 | Yaws et al. (1999) | ? | 21 |
| | $2.6\times10^{-4}$ | | Abraham and Weathersby (1994) | ? | 21 |
| | $3.3\times10^{-4}$ | 2400 | Dean and Lange (1999) | ? | 385, 23 |
| | $4.5\times10^{-4}$ | | Yaws and Yang (1992) | ? | 21 |
| | $3.4\times10^{-4}$ | 2400 | Seinfeld (1986) | ? | 21 |
| | $3.3\times10^{-4}$ | 2400 | Hoffmann and Jacob (1984) | ? | 21 |
| carbon suboxide $C_3O_2$ [504-64-3] GNEVIACKFGQMHB-UHFFFAOYSA-N | $1.1\times10^{-2}$ | | Keßel et al. (2017) | M | 386 |





### A3.2   Alcohols (ROH)

Table A3.2: Alcohols (ROH)

| Substance Formula (Trivial Name) [CAS Registry Number] InChIKey | $H_s^{cp}$ (at $T^\ominus$) $\left[\dfrac{\mathrm{mol}}{\mathrm{m}^3\,\mathrm{Pa}}\right]$ | $\dfrac{\mathrm{d}\ln H_s^{cp}}{\mathrm{d}(1/T)}$ [K] | Reference | Type | Note |
|---|---|---|---|---|---|
| methanol | 2.0 | 5500 | Burkholder et al. (2019) | L | 1 |
| CH$_3$OH | 2.0 | 5500 | Burkholder et al. (2015) | L | 1 |
| [67-56-1] | 2.0 | 5400 | Brockbank (2013) | L | 1 |
| OKKJLVBELUTLKV-UHFFFAOYSA-N | 2.0 | 5600 | Sander et al. (2011) | L | 387, 1 |
| | 2.1 | 5300 | Warneck (2006) | L | |
| | 2.2 | 5200 | Sander et al. (2006) | L | |
| | 2.0 | 5500 | Dohnal et al. (2006) | L | 1 |
| | 1.7 | 4500 | Fogg and Sangster (2003) | L | |
| | 2.1 | 5400 | Plyasunov and Shock (2000) | L | |
| | $3.6\times10^{-2}$ | | St-Pierre et al. (2014) | M | 173 |
| | 2.2 | 5300 | O'Farrell and Waghorne (2010) | M | |
| | 2.1 | | Vitenberg and Dobryakov (2008) | M | |
| | $7.8\times10^{-1}$ | | Helburn et al. (2008) | M | |
| | 2.2 | 5300 | Lin and Chou (2006) | M | |
| | 2.0 | 5600 | Teja et al. (2001) | M | 11, 338 |
| | 2.6 | 5900 | Zhu et al. (2000) | M | |
| | 2.0 | 5500 | Gupta et al. (2000) | M | |
| | 1.6 | | Altschuh et al. (1999) | M | |
| | 2.1 | | Merk and Riederer (1997) | M | |
| | 1.3 | | Kaneko et al. (1994) | M | 14 |
| | 2.2 | | Li and Carr (1993) | M | |
| | 2.6 | 3900 | Pividal et al. (1992) | M | |
| | 2.2 | 5200 | Snider and Dawson (1985) | M | |
| | 2.0 | | Richon et al. (1985) | M | |
| | $1.3\times10^1$ | | Mazza (1980) | M | |
| | 2.2 | | Rytting et al. (1978) | M | |
| | 2.3 | | Burnett (1963) | M | |
| | 2.3 | | Butler et al. (1935) | M | 388 |
| | $7.6\times10^{-2}$ | | Abraham and Acree (2007) | V | |
| | 1.8 | 6200 | Fukuchi et al. (2002) | V | |
| | 1.9 | | Hwang et al. (1992) | V | |
| | 2.8 | | Riederer (1990) | V | |
| | | 5400 | Abraham (1984) | V | |
| | 2.2 | 5700 | Glew and Moelwyn-Hughes (1953) | R | |
| | 2.1 | 5400 | Plyasunov et al. (2001) | T | |
| | 1.5 | | Yaws (2003) | X | 258 |
| | 1.5 | | Yaws (2003) | X | 237 |
| | 1.6 | 5600 | Schaffer and Daubert (1969) | X | 298 |
| | 2.2 | | Gaffney and Senum (1984) | X | 389 |
| | 2.1 | | Timmermans (1960) | X | 390 |
| | 1.6 | | Dupeux et al. (2022) | Q | 259 |
| | 2.1 | | Hayer et al. (2022) | Q | 20 |
| | $5.0\times10^{-1}$ | | Keshavarz et al. (2022) | Q | |
| | 2.8 | | Duchowicz et al. (2020) | Q | 184 |
| | $3.4\times10^{-1}$ | | Wang et al. (2017) | Q | 80, 238 |



Table A3.2: Alcohols (ROH) (...continued)

| Substance<br>Formula<br>(Trivial Name)<br>[CAS Registry Number]<br><small>InChIKey</small> | $H_s^{cp}$<br>(at $T^\ominus$)<br>$\left[\dfrac{\mathrm{mol}}{\mathrm{m^3\,Pa}}\right]$ | $\dfrac{\mathrm{d}\ln H_s^{cp}}{\mathrm{d}(1/T)}$<br><br>[K] | Reference | Type | Note |
|---|---|---|---|---|---|
| | 2.1 | | Wang et al. (2017) | Q | 80, 239 |
| | 6.6 | | Wang et al. (2017) | Q | 80, 240 |
| | 2.1 | | Li et al. (2014) | Q | 241 |
| | 2.0 | | Raventos-Duran et al. (2010) | Q | 242, 243 |
| | $9.9\times10^{-1}$ | | Raventos-Duran et al. (2010) | Q | 244 |
| | 2.5 | | Raventos-Duran et al. (2010) | Q | 245 |
| | $2.6\times10^{-1}$ | | Gharagheizi et al. (2010) | Q | 246 |
| | 2.0 | | Hilal et al. (2008) | Q | |
| | 2.9 | | Modarresi et al. (2007) | Q | 67 |
| | | 6200 | Kühne et al. (2005) | Q | |
| | 2.2 | | Yaffe et al. (2003) | Q | 248, 249 |
| | 1.1 | | Yao et al. (2002) | Q | 229, 267 |
| | 2.6 | | English and Carroll (2001) | Q | 230, 231 |
| | 1.6 | | Katritzky et al. (1998) | Q | |
| | 1.5 | | Yaws et al. (1997) | Q | |
| | 2.0 | | Suzuki et al. (1992) | Q | 232 |
| | 1.8 | | Nirmalakhandan and Speece (1988) | Q | |
| | 2.4 | | Taft et al. (1985) | Q | |
| | 2.2 | | Duchowicz et al. (2020) | ? | 185, 21 |
| | | 5000 | Kühne et al. (2005) | ? | |
| | 1.9 | | Yaws (1999) | ? | 21 |
| | 1.1 | | Abraham and Weathersby (1994) | ? | 21 |
| | 1.4 | | Yaws and Yang (1992) | ? | 21 |
| | 2.2 | | Abraham et al. (1990) | ? | |
| ethanol | 1.9 | 6400 | Burkholder et al. (2019) | L | 1 |
| $C_2H_5OH$ | 1.9 | 6400 | Burkholder et al. (2015) | L | 1 |
| [64-17-5] | 1.8 | 6300 | Brockbank (2013) | L | 1 |
| <small>LFQSCWFLJHTTHZ-UHFFFAOYSA-N</small> | 1.9 | 6400 | Sander et al. (2011) | L | 1 |
| | 1.9 | 6300 | Warneck (2006) | L | |
| | 2.0 | 6600 | Sander et al. (2006) | L | |
| | 1.8 | 6300 | Dohnal et al. (2006) | L | 1 |
| | 1.7 | 5700 | Fogg and Sangster (2003) | L | |
| | 1.9 | 6300 | Plyasunov and Shock (2000) | L | |
| | 1.9 | 6200 | Dubowski (1979) | L | 1 |
| | 1.8 | 5900 | Willey et al. (2017) | M | |
| | 2.2 | 5500 | O'Farrell and Waghorne (2010) | M | |
| | 1.8 | | Vitenberg and Dobryakov (2008) | M | |
| | 1.9 | 5800 | Falabella et al. (2006) | M | 11, 338 |
| | 1.9 | | Straver and de Loos (2005) | M | |
| | | | Cheng et al. (2004) | M | 328 |
| | 1.1 | | Ueberfeld et al. (2001) | M | |
| | 1.8 | 5800 | Gupta et al. (2000) | M | |
| | 1.3 | | Altschuh et al. (1999) | M | |
| | 1.0 | | Eger et al. (1999) | M | 14 |
| | 1.9 | | Merk and Riederer (1997) | M | |
| | $8.3\times10^{-1}$ | | Kaneko et al. (1994) | M | 14 |
| | 1.9 | | Li and Carr (1993) | M | |



Table A3.2: Alcohols (ROH) (...continued)

| Substance<br>Formula<br>(Trivial Name)<br>[CAS Registry Number]<br>InChIKey | $H_s^{cp}$<br>(at $T^\ominus$)<br>$\left[\dfrac{\mathrm{mol}}{\mathrm{m^3\,Pa}}\right]$ | $\dfrac{\mathrm{d}\ln H_s^{cp}}{\mathrm{d}(1/T)}$<br><br>[K] | Reference | Type | Note |
|---|---|---|---|---|---|
|  | 3.1 | 3900 | Pividal et al. (1992) | M |  |
|  | 1.9 |  | Park et al. (1987) | M |  |
|  | 1.9 | 6600 | Snider and Dawson (1985) | M |  |
|  | 2.0 |  | Richon et al. (1985) | M |  |
|  | 1.9 | 6000 | Jones (1983) | M | 1 |
|  | 6.2 |  | Mazza (1980) | M |  |
|  | 1.9 |  | Rytting et al. (1978) | M |  |
|  | 2.3 |  | Rohrschneider (1973) | M |  |
|  | 2.1 |  | Burnett (1963) | M |  |
|  | 1.9 | 6500 | Harger et al. (1950) | M |  |
|  | 1.9 |  | Butler et al. (1935) | M |  |
|  | $4.7\times10^{-2}$ |  | Abraham and Acree (2007) | V |  |
|  | 1.7 | 6300 | Fukuchi et al. (2002) | V |  |
|  | 1.3 |  | Hwang et al. (1992) | V |  |
|  |  | 6300 | Abraham (1984) | V |  |
|  | 1.9 | 6300 | Plyasunov et al. (2001) | T |  |
|  | 1.4 |  | Yaws (2003) | X | 237 |
|  | 1.5 | 6400 | Schaffer and Daubert (1969) | X | 298 |
|  | 2.0 |  | Gaffney and Senum (1984) | X | 389 |
|  | 1.6 |  | Timmermans (1960) | X | 390 |
|  | 1.7 |  | Hayer et al. (2022) | Q | 20 |
|  | $6.7\times10^{-1}$ |  | Keshavarz et al. (2022) | Q |  |
|  | 1.1 |  | Duchowicz et al. (2020) | Q |  |
|  | $3.1\times10^{-1}$ |  | Wang et al. (2017) | Q | 80, 238 |
|  | 2.0 |  | Wang et al. (2017) | Q | 80, 239 |
|  | 2.3 |  | Wang et al. (2017) | Q | 80, 240 |
|  | 1.6 |  | Raventos-Duran et al. (2010) | Q | 242, 243 |
|  | 1.6 |  | Raventos-Duran et al. (2010) | Q | 244 |
|  | 1.6 |  | Raventos-Duran et al. (2010) | Q | 245 |
|  | 1.3 |  | Gharagheizi et al. (2010) | Q | 246 |
|  | 1.1 |  | Hilal et al. (2008) | Q |  |
|  | 1.8 |  | Modarresi et al. (2007) | Q | 67 |
|  |  | 6500 | Kühne et al. (2005) | Q |  |
|  | 2.0 |  | Yaffe et al. (2003) | Q | 248, 249 |
|  | 1.4 |  | Yao et al. (2002) | Q | 229 |
|  | 1.4 |  | English and Carroll (2001) | Q | 230, 231 |
|  | 1.2 |  | Katritzky et al. (1998) | Q |  |
|  | 1.3 |  | Yaws et al. (1997) | Q |  |
|  | 1.4 |  | Russell et al. (1992) | Q | 279 |
|  | 1.4 |  | Suzuki et al. (1992) | Q | 232 |
|  | 1.6 |  | Nirmalakhandan and Speece (1988) | Q |  |
|  | 2.0 |  | Duchowicz et al. (2020) | ? | 185, 21 |
|  | 1.9 |  | Bartelt-Hunt et al. (2008) | ? | 21 |
|  |  | 6400 | Kühne et al. (2005) | ? |  |
|  | 1.2 |  | Yaws (1999) | ? | 21 |
|  | $8.2\times10^{-1}$ |  | Abraham and Weathersby (1994) | ? | 21 |
|  | 1.2 |  | Yaws and Yang (1992) | ? | 21 |



Table A3.2: Alcohols (ROH) (... continued)

| Substance Formula (Trivial Name) [CAS Registry Number] InChIKey | $H_s^{cp}$ (at $T^\ominus$) $\left[\dfrac{\text{mol}}{\text{m}^3\,\text{Pa}}\right]$ | $\dfrac{\text{d}\ln H_s^{cp}}{\text{d}(1/T)}$ [K] | Reference | Type | Note |
|---|---|---|---|---|---|
| | 1.9 | | Abraham et al. (1990) | ? | |
| 1-propanol | 1.4 | 6900 | Burkholder et al. (2019) | L | 391, 1 |
| C$_3$H$_7$OH | 1.4 | 6900 | Burkholder et al. (2015) | L | 392, 1 |
| [71-23-8] | 1.5 | 7000 | Brockbank (2013) | L | 1, 393 |
| BDERNNFJNOPAEC-UHFFFAOYSA-N | 1.4 | 6900 | Sander et al. (2011) | L | 394, 1 |
| | 1.3 | 7500 | Sander et al. (2006) | L | |
| | 1.4 | 6900 | Dohnal et al. (2006) | L | 1 |
| | 1.4 | 6200 | Fogg and Sangster (2003) | L | |
| | 1.5 | 6900 | Plyasunov and Shock (2000) | L | |
| | 1.5 | | Vitenberg and Dobryakov (2008) | M | |
| | 1.2 | 6200 | Falabella et al. (2006) | M | 11, 338 |
| | 1.5 | | Straver and de Loos (2005) | M | |
| | $3.2\times10^{-1}$ | | van Ruth et al. (2002) | M | 14 |
| | $3.2\times10^{-1}$ | | van Ruth and Villeneuve (2002) | M | 14, 361 |
| | $6.5\times10^{-1}$ | | van Ruth et al. (2001) | M | 14 |
| | 1.2 | 6200 | Gupta et al. (2000) | M | |
| | 2.7 | | Altschuh et al. (1999) | M | |
| | 1.5 | | Merk and Riederer (1997) | M | |
| | $7.2\times10^{-1}$ | | Kaneko et al. (1994) | M | 14 |
| | 1.4 | | Li and Carr (1993) | M | |
| | 1.3 | 7500 | Snider and Dawson (1985) | M | |
| | 1.8 | | Richon et al. (1985) | M | |
| | 3.7 | | Mazza (1980) | M | |
| | 1.5 | | Rytting et al. (1978) | M | |
| | 1.6 | | Burnett (1963) | M | |
| | 1.4 | | Butler et al. (1935) | M | 388 |
| | $3.1\times10^{-2}$ | | Abraham and Acree (2007) | V | |
| | 1.8 | 7700 | Fukuchi et al. (2002) | V | |
| | $8.0\times10^{-2}$ | 4500 | Djerki and Laub (1988) | V | |
| | | 6900 | Abraham (1984) | V | |
| | 1.5 | 6900 | Plyasunov et al. (2001) | T | |
| | 1.2 | | Yaws (2003) | X | 258 |
| | 1.2 | | Dupeux et al. (2022) | Q | 259 |
| | 1.5 | | Hayer et al. (2022) | Q | 20 |
| | $9.0\times10^{-1}$ | | Keshavarz et al. (2022) | Q | |
| | 1.2 | | Duchowicz et al. (2020) | Q | |
| | $2.5\times10^{-1}$ | | Wang et al. (2017) | Q | 80, 238 |
| | 1.3 | | Wang et al. (2017) | Q | 80, 239 |
| | 1.2 | | Wang et al. (2017) | Q | 80, 240 |
| | 1.2 | | Raventos-Duran et al. (2010) | Q | 242, 243 |
| | $7.8\times10^{-1}$ | | Raventos-Duran et al. (2010) | Q | 244 |
| | 1.2 | | Raventos-Duran et al. (2010) | Q | 245 |
| | $7.0\times10^{-1}$ | | Hilal et al. (2008) | Q | |
| | 1.1 | | Modarresi et al. (2007) | Q | 67 |
| | | 6900 | Kühne et al. (2005) | Q | |
| | 1.2 | | Yaffe et al. (2003) | Q | 248, 272 |
| | 1.1 | | Yao et al. (2002) | Q | 229 |



Table A3.2: Alcohols (ROH) (...continued)

| Substance Formula (Trivial Name) [CAS Registry Number] InChIKey | $H_s^{cp}$ (at $T^{\ominus}$) $\left[\dfrac{\text{mol}}{\text{m}^3\,\text{Pa}}\right]$ | $\dfrac{\mathrm{d}\ln H_s^{cp}}{\mathrm{d}(1/T)}$ [K] | Reference | Type | Note |
|---|---|---|---|---|---|
| | 1.2 | | English and Carroll (2001) | Q | 230, 231 |
| | 1.7 | | Katritzky et al. (1998) | Q | |
| | 1.2 | | Yaws et al. (1997) | Q | |
| | 1.1 | | Russell et al. (1992) | Q | 279 |
| | 1.1 | | Suzuki et al. (1992) | Q | 232 |
| | 1.2 | | Nirmalakhandan and Speece (1988) | Q | |
| | 1.3 | | Duchowicz et al. (2020) | ? | 185, 21 |
| | | 7500 | Kühne et al. (2005) | ? | |
| | 1.3 | | Yaws (1999) | ? | 21 |
| | $6.0\times10^{-1}$ | | Abraham and Weathersby (1994) | ? | 21 |
| | 1.1 | | Yaws and Yang (1992) | ? | 21 |
| | 1.5 | | Abraham et al. (1990) | ? | |
| 2-propanol C$_3$H$_7$OH (isopropanol) [67-63-0] KFZMGEQAYNKOFK-UHFFFAOYSA-N | 1.2 | 7100 | Burkholder et al. (2019) | L | 1 |
| | 1.2 | 7100 | Burkholder et al. (2015) | L | 1 |
| | 1.2 | 6900 | Brockbank (2013) | L | 1 |
| | 1.3 | 7500 | Sander et al. (2011) | L | |
| | 1.3 | 7500 | Sander et al. (2006) | L | |
| | 1.2 | 6200 | Fogg and Sangster (2003) | L | |
| | 1.2 | 7000 | Plyasunov and Shock (2000) | L | |
| | 1.1 | 8400 | Hiatt (2013) | M | |
| | $6.8\times10^{-1}$ | | Helburn et al. (2008) | M | |
| | 1.3 | 7300 | Lin and Chou (2006) | M | |
| | | | Cheng et al. (2004) | M | 328 |
| | | | Cheng et al. (2003) | M | 328 |
| | $1.8\times10^{-1}$ | | Ayuttaya et al. (2001) | M | 340 |
| | $1.0\times10^{-3}$ | | Ayuttaya et al. (2001) | M | 341 |
| | $5.7\times10^{-1}$ | | Ayuttaya et al. (2001) | M | 342 |
| | 1.1 | | Kim et al. (2000) | M | |
| | $9.2\times10^{-1}$ | | Altschuh et al. (1999) | M | |
| | 1.2 | | Merk and Riederer (1997) | M | |
| | $5.8\times10^{-1}$ | | Kaneko et al. (1994) | M | 14 |
| | $7.9\times10^{-1}$ | 5700 | Kolb et al. (1992) | M | 277 |
| | 1.4 | | Pividal et al. (1992) | M | 80 |
| | $9.8\times10^{-1}$ | | Yu (1992) | M | 12 |
| | 1.2 | 7400 | Snider and Dawson (1985) | M | |
| | 2.1 | | Mazza (1980) | M | |
| | 1.2 | | Rytting et al. (1978) | M | |
| | 1.2 | | Butler et al. (1935) | M | |
| | 1.2 | 7100 | Fenclová et al. (2007) | V | 1 |
| | 1.2 | 7600 | Fukuchi et al. (2002) | V | |
| | 1.7 | | Hine and Weimar (1965) | R | |
| | $7.6\times10^{-1}$ | | Yaws (2003) | X | 258 |
| | 2.4 | | Dupeux et al. (2022) | Q | 259 |
| | 1.2 | | Hayer et al. (2022) | Q | 20 |
| | $9.0\times10^{-1}$ | | Keshavarz et al. (2022) | Q | |
| | $4.8\times10^{-1}$ | | Duchowicz et al. (2020) | Q | 184 |
| | $2.8\times10^{-1}$ | | Wang et al. (2017) | Q | 80, 238 |





Table A3.2: Alcohols (ROH) (...continued)

| Substance<br>Formula<br>(Trivial Name)<br>[CAS Registry Number]<br><small>InChIKey</small> | $H_s^{cp}$<br>(at $T^\ominus$)<br>$\left[\dfrac{\mathrm{mol}}{\mathrm{m^3\,Pa}}\right]$ | $\dfrac{\mathrm{d}\ln H_s^{cp}}{\mathrm{d}(1/T)}$<br><br>[K] | Reference | Type | Note |
|---|---|---|---|---|---|
| | 1.0 | | Wang et al. (2017) | Q | 80, 239 |
| | 1.2 | | Wang et al. (2017) | Q | 80, 240 |
| | 1.2 | | Raventos-Duran et al. (2010) | Q | 242, 243 |
| | $6.2\times10^{-1}$ | | Raventos-Duran et al. (2010) | Q | 244 |
| | 1.2 | | Raventos-Duran et al. (2010) | Q | 245 |
| | $4.3\times10^{-1}$ | | Hilal et al. (2008) | Q | |
| | $7.0\times10^{-1}$ | | Modarresi et al. (2007) | Q | 67 |
| | | 6900 | Kühne et al. (2005) | Q | |
| | | | Yaffe et al. (2003) | Q | 356 |
| | $7.7\times10^{-1}$ | | Yao et al. (2002) | Q | 229 |
| | 1.0 | | English and Carroll (2001) | Q | 230, 274 |
| | 1.1 | | Katritzky et al. (1998) | Q | |
| | $8.9\times10^{-1}$ | | Yaws et al. (1997) | Q | |
| | 1.4 | | Russell et al. (1992) | Q | 279 |
| | $9.7\times10^{-1}$ | | Suzuki et al. (1992) | Q | 232 |
| | 1.1 | | Nirmalakhandan and Speece (1988) | Q | |
| | 1.3 | | Taft et al. (1985) | Q | |
| | 1.2 | | Duchowicz et al. (2020) | ? | 185, 21 |
| | | 6000 | Kühne et al. (2005) | ? | |
| | $8.0\times10^{-1}$ | | Yaws (1999) | ? | 21 |
| | $5.0\times10^{-1}$ | | Abraham and Weathersby (1994) | ? | 21 |
| | $8.8\times10^{-1}$ | | Yaws and Yang (1992) | ? | 21 |
| | 1.2 | | Abraham et al. (1990) | ? | |
| glycidol<br>$C_3H_6O_2$<br>[556-52-5]<br><small>CTKINSOISVBQLD-UHFFFAOYSA-N</small> | $1.7\times10^3$ | | HSDB (2015) | Q | 99 |
| 1-butanol<br>$C_4H_9OH$<br>[71-36-3]<br><small>LRHPLDYGYMQRHN-UHFFFAOYSA-N</small> | 1.2 | 7500 | Burkholder et al. (2019) | L | 1 |
| | 1.2 | 7500 | Burkholder et al. (2015) | L | 1 |
| | 1.2 | 7500 | Brockbank (2013) | L | 1 |
| | 1.2 | 7500 | Sander et al. (2011) | L | 1 |
| | 1.3 | 7200 | Sander et al. (2006) | L | |
| | 1.2 | 7500 | Dohnal et al. (2006) | L | 1 |
| | 1.1 | 6300 | Fogg and Sangster (2003) | L | |
| | 1.2 | 7400 | Plyasunov and Shock (2000) | L | |
| | 1.0 | 7000 | Wu et al. (2022a) | M | |
| | 2.0 | | Chao et al. (2017) | M | |
| | 1.0 | 6800 | Shunthirasingham et al. (2013) | M | |
| | 1.3 | | Vitenberg and Dobryakov (2008) | M | |
| | 1.1 | 6000 | Lei et al. (2007) | M | 395 |
| | $8.2\times10^{-1}$ | 6200 | Falabella et al. (2006) | M | 11, 338 |
| | $9.4\times10^{-1}$ | 6100 | Hovorka et al. (2002) | M | 11 |
| | $4.5\times10^{-1}$ | | van Ruth et al. (2002) | M | 14 |
| | $4.4\times10^{-1}$ | | van Ruth and Villeneuve (2002) | M | 14, 361 |
| | $4.8\times10^{-1}$ | | van Ruth et al. (2001) | M | 14 |
| | 1.1 | | Kim et al. (2000) | M | |





Table A3.2: Alcohols (ROH) (... continued)

| Substance Formula (Trivial Name) [CAS Registry Number] InChIKey | $H_s^{cp}$ (at $T^{\ominus}$) $\left[\dfrac{\text{mol}}{\text{m}^3\,\text{Pa}}\right]$ | $\dfrac{\text{d}\ln H_s^{cp}}{\text{d}(1/T)}$ [K] | Reference | Type | Note |
|---|---|---|---|---|---|
| | $8.2\times10^{-1}$ | 6200 | Gupta et al. (2000) | M | |
| | 1.2 | | Altschuh et al. (1999) | M | |
| | $5.5\times10^{-1}$ | | Eger et al. (1999) | M | 14 |
| | 1.1 | | Merk and Riederer (1997) | M | |
| | $1.4\times10^{-1}$ | | Chaintreau et al. (1995) | M | |
| | $5.1\times10^{-1}$ | | Kaneko et al. (1994) | M | 14 |
| | 1.1 | | Li and Carr (1993) | M | |
| | $6.1\times10^{-1}$ | 5600 | Kolb et al. (1992) | M | 277 |
| | 1.2 | 7200 | Snider and Dawson (1985) | M | |
| | $5.3\times10^{-1}$ | | Friant and Suffet (1979) | M | 38 |
| | 1.2 | | Rytting et al. (1978) | M | |
| | 1.1 | | Amoore and Buttery (1978) | M | |
| | 1.1 | | Buttery et al. (1969) | M | |
| | 1.4 | | Burnett (1963) | M | |
| | 1.2 | | Butler et al. (1935) | M | 388 |
| | 1.1 | | Chao et al. (2017) | V | |
| | 1.1 | | Mackay et al. (2006c) | V | |
| | $7.3\times10^{-1}$ | | Mackay et al. (1995) | V | |
| | $8.3\times10^{-1}$ | | Hwang et al. (1992) | V | |
| | $2.2\times10^{-1}$ | 4700 | Djerki and Laub (1988) | V | |
| | | 7400 | Abraham (1984) | V | |
| | 1.2 | | Amoore and Buttery (1978) | V | |
| | 1.2 | | Butler et al. (1935) | V | |
| | 1.2 | | Yaws (2003) | X | 258 |
| | $9.4\times10^{-1}$ | | Dupeux et al. (2022) | Q | 259 |
| | 1.9 | | Hayer et al. (2022) | Q | 20 |
| | 1.2 | | Keshavarz et al. (2022) | Q | |
| | 1.3 | | Duchowicz et al. (2020) | Q | 184 |
| | $2.0\times10^{-1}$ | | Wang et al. (2017) | Q | 80, 238 |
| | $9.1\times10^{-1}$ | | Wang et al. (2017) | Q | 80, 239 |
| | 1.3 | | Wang et al. (2017) | Q | 80, 240 |
| | 1.2 | | Li et al. (2014) | Q | 241 |
| | $9.4\times10^{-1}$ | | Gharagheizi et al. (2012) | Q | |
| | $7.8\times10^{-1}$ | | Raventos-Duran et al. (2010) | Q | 242, 243 |
| | $6.2\times10^{-1}$ | | Raventos-Duran et al. (2010) | Q | 244 |
| | $9.9\times10^{-1}$ | | Raventos-Duran et al. (2010) | Q | 245 |
| | $5.6\times10^{-1}$ | | Hilal et al. (2008) | Q | |
| | 1.0 | | Modarresi et al. (2007) | Q | 67 |
| | | 7200 | Kühne et al. (2005) | Q | |
| | 1.2 | | Yaffe et al. (2003) | Q | 248, 272 |
| | 1.1 | | Yao et al. (2002) | Q | 229 |
| | $9.5\times10^{-1}$ | | English and Carroll (2001) | Q | 230, 260 |
| | 1.8 | | Katritzky et al. (1998) | Q | |
| | 1.1 | | Yaws et al. (1997) | Q | |
| | $8.6\times10^{-1}$ | | Russell et al. (1992) | Q | 279 |
| | $8.4\times10^{-1}$ | | Suzuki et al. (1992) | Q | 232 |
| | $9.9\times10^{-1}$ | | Nirmalakhandan and Speece (1988) | Q | |



Table A3.2: Alcohols (ROH) (...continued)

| Substance Formula (Trivial Name) [CAS Registry Number] InChIKey | $H_s^{cp}$ (at $T^{\ominus}$) $\left[\dfrac{\text{mol}}{\text{m}^3\,\text{Pa}}\right]$ | $\dfrac{\text{d}\ln H_s^{cp}}{\text{d}(1/T)}$ [K] | Reference | Type | Note |
|---|---|---|---|---|---|
| | 1.1 | | Duchowicz et al. (2020) | ? | 185, 21 |
| | | 6900 | Kühne et al. (2005) | ? | |
| | 1.1 | | Yaws (1999) | ? | 21 |
| | 1.2 | | Abraham et al. (1990) | ? | |
| | 1.8 | | Mackay and Yeun (1983) | ? | |
| 2-butanol | 1.1 | 7600 | Burkholder et al. (2019) | L | 1 |
| $C_4H_{10}O$ | 1.1 | 7600 | Burkholder et al. (2015) | L | 1 |
| (*sec*-butanol) | $9.5\times10^{-1}$ | 7600 | Brockbank (2013) | L | 1 |
| [78-92-2] | 1.1 | 7300 | Sander et al. (2011) | L | |
| BTANRVKWQNVYAZ-UHFFFAOYSA-N | 1.1 | 7300 | Sander et al. (2006) | L | |
| | 1.0 | 7400 | Fogg and Sangster (2003) | L | |
| | $9.9\times10^{-1}$ | 7600 | Plyasunov and Shock (2000) | L | |
| | $8.3\times10^{-1}$ | | Merk and Riederer (1997) | M | |
| | 1.1 | 7300 | Snider and Dawson (1985) | M | |
| | $9.8\times10^{-1}$ | | Rytting et al. (1978) | M | |
| | $9.6\times10^{-1}$ | | Butler et al. (1935) | M | |
| | $9.5\times10^{-1}$ | 7600 | Fenclová et al. (2007) | V | 1 |
| | 1.1 | | Mackay et al. (2006c) | V | |
| | 1.1 | | Mackay et al. (1995) | V | |
| | $9.1\times10^{-1}$ | 7500 | Cabani et al. (1975b) | T | |
| | $6.7\times10^{-1}$ | | Yaws (2003) | X | 258 |
| | 1.1 | | Dupeux et al. (2022) | Q | 259 |
| | 1.2 | | Hayer et al. (2022) | Q | 20 |
| | 1.2 | | Keshavarz et al. (2022) | Q | |
| | $5.1\times10^{-1}$ | | Duchowicz et al. (2020) | Q | 299 |
| | $2.3\times10^{-1}$ | | Wang et al. (2017) | Q | 80, 238 |
| | $8.9\times10^{-1}$ | | Wang et al. (2017) | Q | 80, 239 |
| | $7.3\times10^{-1}$ | | Wang et al. (2017) | Q | 80, 240 |
| | $7.8\times10^{-1}$ | | Raventos-Duran et al. (2010) | Q | 242, 243 |
| | $4.9\times10^{-1}$ | | Raventos-Duran et al. (2010) | Q | 244 |
| | $9.9\times10^{-1}$ | | Raventos-Duran et al. (2010) | Q | 245 |
| | $3.9\times10^{-1}$ | | Hilal et al. (2008) | Q | |
| | $4.9\times10^{-1}$ | | Modarresi et al. (2007) | Q | 67 |
| | | 7200 | Kühne et al. (2005) | Q | |
| | 1.2 | | Yaffe et al. (2003) | Q | 248, 249 |
| | $6.2\times10^{-1}$ | | Yao et al. (2002) | Q | 229 |
| | $8.4\times10^{-1}$ | | English and Carroll (2001) | Q | 230, 231 |
| | $5.7\times10^{-1}$ | | Katritzky et al. (1998) | Q | |
| | 1.2 | | Yaws et al. (1997) | Q | |
| | $7.7\times10^{-1}$ | | Suzuki et al. (1992) | Q | 232 |
| | $9.0\times10^{-1}$ | | Nirmalakhandan and Speece (1988) | Q | |
| | 1.1 | | Duchowicz et al. (2020) | ? | 185, 21 |
| | | 7100 | Kühne et al. (2005) | ? | |
| | 1.2 | | Yaws (1999) | ? | 21 |
| | $9.9\times10^{-1}$ | | Abraham et al. (1990) | ? | |



Table A3.2: Alcohols (ROH) (... continued)

| Substance Formula (Trivial Name) [CAS Registry Number] InChIKey | $H_s^{cp}$ (at $T^{\ominus}$) $\left[\dfrac{\text{mol}}{\text{m}^3\,\text{Pa}}\right]$ | $\dfrac{\text{d}\ln H_s^{cp}}{\text{d}(1/T)}$ [K] | Reference | Type | Note |
|---|---|---|---|---|---|
| 2-methyl-1-propanol | $8.3\times10^{-1}$ | 7200 | Burkholder et al. (2019) | L | 1 |
| $C_4H_{10}O$ | $8.3\times10^{-1}$ | 7200 | Burkholder et al. (2015) | L | 1 |
| (isobutanol) | $8.0\times10^{-1}$ | 7400 | Brockbank (2013) | L | 1 |
| [78-83-1] | 1.0 | | Sander et al. (2011) | L | |
| ZXEKIIBDNHEJCQ-UHFFFAOYSA-N | 1.0 | | Sander et al. (2006) | L | |
| | $8.5\times10^{-1}$ | 7200 | Plyasunov and Shock (2000) | L | |
| | 1.2 | | Chao et al. (2017) | M | |
| | $2.2\times10^{-1}$ | | Kim and Kim (2014) | M | |
| | | | Cheng et al. (2004) | M | 328 |
| | $6.7\times10^{-1}$ | 6100 | Hovorka et al. (2002) | M | 11 |
| | 1.1 | | Altschuh et al. (1999) | M | |
| | $3.7\times10^{-1}$ | | Shiu and Mackay (1997) | M | |
| | $7.8\times10^{-1}$ | | Merk and Riederer (1997) | M | |
| | $4.4\times10^{-1}$ | | Kaneko et al. (1994) | M | 14 |
| | 1.0 | | Snider and Dawson (1985) | M | |
| | $8.0\times10^{-1}$ | | Rytting et al. (1978) | M | |
| | $8.3\times10^{-1}$ | | Butler et al. (1935) | M | |
| | $9.2\times10^{-1}$ | | Chao et al. (2017) | V | |
| | $7.6\times10^{-1}$ | 7200 | Fenclová et al. (2007) | V | 1 |
| | $7.3\times10^{-1}$ | | Mackay et al. (2006c) | V | |
| | $7.3\times10^{-1}$ | | Shiu and Mackay (1997) | V | |
| | $7.3\times10^{-1}$ | | Mackay et al. (1995) | V | |
| | 1.2 | | Keshavarz et al. (2022) | Q | |
| | $5.1\times10^{-1}$ | | Duchowicz et al. (2020) | Q | |
| | $2.3\times10^{-1}$ | | Wang et al. (2017) | Q | 80, 238 |
| | $9.1\times10^{-1}$ | | Wang et al. (2017) | Q | 80, 239 |
| | $7.6\times10^{-1}$ | | Wang et al. (2017) | Q | 80, 240 |
| | $7.8\times10^{-1}$ | | Raventos-Duran et al. (2010) | Q | 242, 243 |
| | $6.2\times10^{-1}$ | | Raventos-Duran et al. (2010) | Q | 244 |
| | $9.9\times10^{-1}$ | | Raventos-Duran et al. (2010) | Q | 245 |
| | $5.1\times10^{-1}$ | | Hilal et al. (2008) | Q | |
| | $6.1\times10^{-1}$ | | Modarresi et al. (2007) | Q | 67 |
| | | 7200 | Kühne et al. (2005) | Q | |
| | $3.7\times10^{-1}$ | | Yaffe et al. (2003) | Q | 248, 249 |
| | $6.2\times10^{-1}$ | | Yao et al. (2002) | Q | 229 |
| | $5.7\times10^{-1}$ | | English and Carroll (2001) | Q | 230, 231 |
| | 1.8 | | Katritzky et al. (1998) | Q | |
| | $8.3\times10^{-1}$ | | Yaws et al. (1997) | Q | |
| | $7.7\times10^{-1}$ | | Suzuki et al. (1992) | Q | 232 |
| | $8.4\times10^{-1}$ | | Nirmalakhandan and Speece (1988) | Q | |
| | 1.0 | | Duchowicz et al. (2020) | ? | 185, 21 |
| | | 8100 | Kühne et al. (2005) | ? | |
| | $8.4\times10^{-1}$ | | Yaws (1999) | ? | 21 |
| | $3.1\times10^{-1}$ | | Abraham and Weathersby (1994) | ? | 21 |
| | $8.0\times10^{-1}$ | | Abraham et al. (1990) | ? | |
| | $9.6\times10^{-1}$ | | Mackay and Yeun (1983) | ? | |



Table A3.2: Alcohols (ROH) (...continued)

| Substance Formula (Trivial Name) [CAS Registry Number] InChIKey | $H_s^{cp}$ (at $T^\ominus$) $\left[\dfrac{\text{mol}}{\text{m}^3\,\text{Pa}}\right]$ | $\dfrac{\text{d}\ln H_s^{cp}}{\text{d}(1/T)}$ [K] | Reference | Type | Note |
|---|---|---|---|---|---|
| 2-methyl-2-propanol | $7.9\times10^{-1}$ | 7600 | Brockbank (2013) | L | 1 |
| C$_4$H$_{10}$O | $6.9\times10^{-1}$ | 8300 | Sander et al. (2011) | L | |
| (*tert*-butanol) | $6.9\times10^{-1}$ | 8300 | Sander et al. (2006) | L | |
| [75-65-0] | $7.9\times10^{-1}$ | 7700 | Plyasunov and Shock (2000) | L | |
| DKGAVHZHDRPRBM-UHFFFAOYSA-N | 1.4 | 7900 | Hiatt (2013) | M | |
| | 1.1 | | Altschuh et al. (1999) | M | |
| | $8.1\times10^{-1}$ | | Merk and Riederer (1997) | M | |
| | | | Koga (1995) | M | 396 |
| | $6.8\times10^{-1}$ | 8300 | Snider and Dawson (1985) | M | |
| | $7.6\times10^{-1}$ | | Rytting et al. (1978) | M | |
| | $8.3\times10^{-1}$ | | Butler et al. (1935) | M | |
| | $8.0\times10^{-1}$ | 7700 | Fenclová et al. (2007) | V | 1 |
| | $2.4\times10^{-1}$ | | Yaws (2003) | X | 258 |
| | $8.0\times10^{-1}$ | 6500 | Pankow et al. (1996) | C | |
| | 2.0 | | Dupeux et al. (2022) | Q | 259 |
| | 1.2 | | Keshavarz et al. (2022) | Q | |
| | $2.2\times10^{-1}$ | | Duchowicz et al. (2020) | Q | 184 |
| | $1.6\times10^{-1}$ | | Wang et al. (2017) | Q | 80, 238 |
| | $6.6\times10^{-1}$ | | Wang et al. (2017) | Q | 80, 239 |
| | 1.1 | | Wang et al. (2017) | Q | 80, 240 |
| | $7.8\times10^{-1}$ | | Raventos-Duran et al. (2010) | Q | 242, 243 |
| | $3.1\times10^{-1}$ | | Raventos-Duran et al. (2010) | Q | 244 |
| | $9.9\times10^{-1}$ | | Raventos-Duran et al. (2010) | Q | 245 |
| | $2.2\times10^{-1}$ | | Hilal et al. (2008) | Q | |
| | $4.1\times10^{-1}$ | | Modarresi et al. (2007) | Q | 67 |
| | | 7200 | Kühne et al. (2005) | Q | |
| | $7.9\times10^{-1}$ | | Yaffe et al. (2003) | Q | 248, 249 |
| | $2.9\times10^{-1}$ | | Yao et al. (2002) | Q | 229 |
| | $9.0\times10^{-1}$ | | English and Carroll (2001) | Q | 230, 231 |
| | $7.5\times10^{-1}$ | | Katritzky et al. (1998) | Q | |
| | $7.3\times10^{-1}$ | | Nirmalakhandan et al. (1997) | Q | |
| | $3.0\times10^{-1}$ | | Yaws et al. (1997) | Q | |
| | $6.1\times10^{-1}$ | | Suzuki et al. (1992) | Q | 232 |
| | $7.0\times10^{-1}$ | | Nirmalakhandan and Speece (1988) | Q | |
| | 1.1 | | Duchowicz et al. (2020) | ? | 185, 21 |
| | | 8300 | Kühne et al. (2005) | ? | |
| | $5.7\times10^{-1}$ | | Yaws (1999) | ? | 21 |
| | $7.7\times10^{-1}$ | | Abraham et al. (1990) | ? | |
| | | | Burkholder et al. (2019) | W | 397 |
| | | | Burkholder et al. (2015) | W | 398 |
| 1-pentanol | $8.6\times10^{-1}$ | 7400 | Brockbank (2013) | L | 1 |
| C$_5$H$_{11}$OH | 1.0 | 7900 | Dohnal et al. (2006) | L | 1 |
| (amyl alcohol) | $8.9\times10^{-1}$ | 7800 | Plyasunov and Shock (2000) | L | |
| [71-41-0] | $8.1\times10^{-1}$ | 7100 | Shunthirasingham et al. (2013) | M | |
| AMQJEAYHLZJPGS-UHFFFAOYSA-N | $7.5\times10^{-1}$ | 6100 | Lei et al. (2007) | M | 395 |
| | $9.4\times10^{-1}$ | 6800 | Falabella et al. (2006) | M | 11, 338 |
| | $9.5\times10^{-1}$ | 6900 | Gupta et al. (2000) | M | |





Table A3.2: Alcohols (ROH) (...continued)

| Substance Formula (Trivial Name) [CAS Registry Number] InChIKey | $H_s^{cp}$ (at $T^\ominus$) $\left[\dfrac{\text{mol}}{\text{m}^3\,\text{Pa}}\right]$ | $\dfrac{\text{d}\ln H_s^{cp}}{\text{d}(1/T)}$ [K] | Reference | Type | Note |
|---|---|---|---|---|---|
| | $7.7 \times 10^{-1}$ | | Merk and Riederer (1997) | M | |
| | $4.2 \times 10^{-1}$ | | Kaneko et al. (1994) | M | 14 |
| | $8.4 \times 10^{-1}$ | | Li and Carr (1993) | M | |
| | $9.0 \times 10^{-1}$ | | Rytting et al. (1978) | M | |
| | $7.8 \times 10^{-1}$ | | Butler et al. (1935) | M | |
| | $8.3 \times 10^{-1}$ | | Mackay et al. (2006c) | V | |
| | $8.3 \times 10^{-1}$ | | Mackay et al. (1995) | V | |
| | $6.1 \times 10^{-1}$ | 4900 | Djerki and Laub (1988) | V | |
| | | 7800 | Abraham (1984) | V | |
| | $7.8 \times 10^{-1}$ | | Amoore and Buttery (1978) | V | |
| | $7.6 \times 10^{-1}$ | | Butler et al. (1935) | V | |
| | $7.6 \times 10^{-1}$ | | Yaws (2003) | X | 258 |
| | $7.3 \times 10^{-1}$ | | Dupeux et al. (2022) | Q | 259 |
| | 1.6 | | Keshavarz et al. (2022) | Q | |
| | 1.4 | | Duchowicz et al. (2020) | Q | |
| | $1.8 \times 10^{-1}$ | | Wang et al. (2017) | Q | 80, 238 |
| | $6.6 \times 10^{-1}$ | | Wang et al. (2017) | Q | 80, 239 |
| | $9.1 \times 10^{-1}$ | | Wang et al. (2017) | Q | 80, 240 |
| | $6.2 \times 10^{-1}$ | | Raventos-Duran et al. (2010) | Q | 242, 243 |
| | $3.9 \times 10^{-1}$ | | Raventos-Duran et al. (2010) | Q | 244 |
| | $7.8 \times 10^{-1}$ | | Raventos-Duran et al. (2010) | Q | 245 |
| | $4.5 \times 10^{-1}$ | | Hilal et al. (2008) | Q | |
| | $7.6 \times 10^{-1}$ | | Modarresi et al. (2007) | Q | 67 |
| | | 7600 | Kühne et al. (2005) | Q | |
| | $7.9 \times 10^{-1}$ | | Yaffe et al. (2003) | Q | 248, 249 |
| | $5.3 \times 10^{-1}$ | | Yao et al. (2002) | Q | 229 |
| | $7.7 \times 10^{-1}$ | | English and Carroll (2001) | Q | 230, 231 |
| | 1.3 | | Katritzky et al. (1998) | Q | |
| | $7.7 \times 10^{-1}$ | | Yaws et al. (1997) | Q | |
| | $6.2 \times 10^{-1}$ | | Russell et al. (1992) | Q | 358 |
| | $6.5 \times 10^{-1}$ | | Suzuki et al. (1992) | Q | 232 |
| | $7.9 \times 10^{-1}$ | | Nirmalakhandan and Speece (1988) | Q | |
| | $7.6 \times 10^{-1}$ | | Duchowicz et al. (2020) | ? | 185, 21 |
| | | 7700 | Kühne et al. (2005) | ? | |
| | $7.7 \times 10^{-1}$ | | Yaws (1999) | ? | 21 |
| | $8.1 \times 10^{-1}$ | | Yaws and Yang (1992) | ? | 21 |
| | $9.0 \times 10^{-1}$ | | Abraham et al. (1990) | ? | |
| | $9.6 \times 10^{-1}$ | | Mackay and Yeun (1983) | ? | |
| 2-pentanol $C_5H_{12}O$ (*sec*-pentanol) [6032-29-7] JYVLIDXNZAXMDK-UHFFFAOYSA-N | $6.7 \times 10^{-1}$ | 7900 | Brockbank (2013) | L | 1 |
| | $6.5 \times 10^{-1}$ | 8000 | Plyasunov and Shock (2000) | L | |
| | $7.1 \times 10^{-1}$ | 8000 | Fenclová et al. (2010) | M | 1 |
| | $3.0 \times 10^{-1}$ | | van Ruth et al. (2002) | M | 14 |
| | $2.6 \times 10^{-1}$ | | van Ruth and Villeneuve (2002) | M | 14, 361 |
| | $2.3 \times 10^{-1}$ | | van Ruth et al. (2001) | M | 14 |
| | $6.6 \times 10^{-1}$ | | Merk and Riederer (1997) | M | |
| | $6.7 \times 10^{-1}$ | | Butler et al. (1935) | M | |
| | $6.6 \times 10^{-1}$ | | Mackay et al. (2006c) | V | |





Table A3.2: Alcohols (ROH) (...continued)

| Substance Formula (Trivial Name) [CAS Registry Number] InChIKey | $H_s^{cp}$ (at $T^{\ominus}$) $\left[\dfrac{\text{mol}}{\text{m}^3\,\text{Pa}}\right]$ | $\dfrac{\text{d}\ln H_s^{cp}}{\text{d}(1/T)}$ [K] | Reference | Type | Note |
|---|---|---|---|---|---|
| | $6.6\times10^{-1}$ | | Mackay et al. (1995) | V | |
| | $6.5\times10^{-1}$ | | Yaws (2003) | X | 258 |
| | $7.6\times10^{-1}$ | | Dupeux et al. (2022) | Q | 259 |
| | 1.6 | | Keshavarz et al. (2022) | Q | |
| | $5.3\times10^{-1}$ | | Duchowicz et al. (2020) | Q | |
| | $1.8\times10^{-1}$ | | Wang et al. (2017) | Q | 80, 238 |
| | $6.3\times10^{-1}$ | | Wang et al. (2017) | Q | 80, 239 |
| | $6.2\times10^{-1}$ | | Wang et al. (2017) | Q | 80, 240 |
| | $6.2\times10^{-1}$ | | Raventos-Duran et al. (2010) | Q | 271, 243 |
| | $3.9\times10^{-1}$ | | Raventos-Duran et al. (2010) | Q | 244 |
| | $7.8\times10^{-1}$ | | Raventos-Duran et al. (2010) | Q | 245 |
| | $3.1\times10^{-1}$ | | Hilal et al. (2008) | Q | |
| | $3.5\times10^{-1}$ | | Modarresi et al. (2007) | Q | 67 |
| | | 7600 | Kühne et al. (2005) | Q | |
| | $6.7\times10^{-1}$ | | Yaffe et al. (2003) | Q | 248, 249 |
| | $6.4\times10^{-1}$ | | Yao et al. (2002) | Q | 229 |
| | $7.0\times10^{-1}$ | | English and Carroll (2001) | Q | 230, 260 |
| | $9.0\times10^{-1}$ | | Katritzky et al. (1998) | Q | |
| | $6.5\times10^{-1}$ | | Yaws et al. (1997) | Q | |
| | $5.8\times10^{-1}$ | | Suzuki et al. (1992) | Q | 232 |
| | $7.2\times10^{-1}$ | | Nirmalakhandan and Speece (1988) | Q | |
| | $6.7\times10^{-1}$ | | Duchowicz et al. (2020) | ? | 185, 21 |
| | | 7900 | Kühne et al. (2005) | ? | |
| | $6.5\times10^{-1}$ | | Yaws (1999) | ? | 21 |
| | $6.7\times10^{-1}$ | | Abraham et al. (1990) | ? | |
| 3-pentanol $C_5H_{12}O$ [584-02-1] AQIXEPGDORPWBJ-UHFFFAOYSA-N | $6.0\times10^{-1}$ | 7600 | Brockbank (2013) | L | 1 |
| | $5.5\times10^{-1}$ | | Plyasunov and Shock (2000) | L | |
| | $6.4\times10^{-1}$ | 8000 | Fenclová et al. (2010) | M | 1 |
| | $5.0\times10^{-1}$ | | Duchowicz et al. (2020) | V | 186 |
| | $6.3\times10^{-1}$ | 7900 | Cabani et al. (1975b) | T | |
| | $5.3\times10^{-1}$ | | Duchowicz et al. (2020) | Q | |
| | $1.8\times10^{-1}$ | | Wang et al. (2017) | Q | 80, 238 |
| | $6.6\times10^{-1}$ | | Wang et al. (2017) | Q | 80, 239 |
| | $2.3\times10^{-1}$ | | Wang et al. (2017) | Q | 80, 240 |
| | $6.2\times10^{-1}$ | | Raventos-Duran et al. (2010) | Q | 242, 243 |
| | $3.9\times10^{-1}$ | | Raventos-Duran et al. (2010) | Q | 244 |
| | $7.8\times10^{-1}$ | | Raventos-Duran et al. (2010) | Q | 245 |
| | $3.2\times10^{-1}$ | | Hilal et al. (2008) | Q | |
| | $5.2\times10^{-1}$ | | Modarresi et al. (2007) | Q | 67 |
| | | 7600 | Kühne et al. (2005) | Q | |
| | $6.7\times10^{-1}$ | | Yaffe et al. (2003) | Q | 248, 272 |
| | $6.1\times10^{-1}$ | | Yao et al. (2002) | Q | 229 |
| | $7.0\times10^{-1}$ | | English and Carroll (2001) | Q | 230, 231 |
| | $5.0\times10^{-1}$ | | Katritzky et al. (1998) | Q | |
| | $7.7\times10^{-1}$ | | Nirmalakhandan et al. (1997) | Q | |
| | $5.2\times10^{-1}$ | | Yaws et al. (1997) | Q | |
| | | 7500 | Kühne et al. (2005) | ? | |



Table A3.2: Alcohols (ROH) (... continued)

| Substance Formula (Trivial Name) [CAS Registry Number] InChIKey | $H_s^{cp}$ (at $T^{\ominus}$) $\left[\dfrac{\mathrm{mol}}{\mathrm{m^3\,Pa}}\right]$ | $\dfrac{\mathrm{d}\ln H_s^{cp}}{\mathrm{d}(1/T)}$ [K] | Reference | Type | Note |
|---|---|---|---|---|---|
| | $5.3\times10^{-1}$ | | Yaws (1999) | ? | 21 |
| | $6.2\times10^{-1}$ | | Abraham et al. (1990) | ? | |
| 2-methyl-1-butanol | $7.6\times10^{-1}$ | 7900 | Brockbank (2013) | L | 1 |
| $C_5H_{12}O$ | $7.1\times10^{-1}$ | | Plyasunov and Shock (2000) | L | |
| [137-32-6] | $7.8\times10^{-1}$ | 7700 | Fenclová et al. (2010) | M | 1 |
| QPRQEDXDYOZYLA-UHFFFAOYSA-N | $3.3\times10^{-1}$ | | Kaneko et al. (1994) | M | 14 |
| | $7.0\times10^{-1}$ | | Butler et al. (1935) | M | |
| | $8.4\times10^{-1}$ | | Yaws (2003) | X | 258 |
| | $4.6\times10^{-1}$ | | Dupeux et al. (2022) | Q | 259 |
| | 1.6 | | Keshavarz et al. (2022) | Q | |
| | $5.3\times10^{-1}$ | | Duchowicz et al. (2020) | Q | 299 |
| | $1.8\times10^{-1}$ | | Wang et al. (2017) | Q | 80, 238 |
| | $7.6\times10^{-1}$ | | Wang et al. (2017) | Q | 80, 239 |
| | $5.5\times10^{-1}$ | | Wang et al. (2017) | Q | 80, 240 |
| | $6.2\times10^{-1}$ | | Raventos-Duran et al. (2010) | Q | 242, 243 |
| | $4.9\times10^{-1}$ | | Raventos-Duran et al. (2010) | Q | 244 |
| | $7.8\times10^{-1}$ | | Raventos-Duran et al. (2010) | Q | 245 |
| | $3.9\times10^{-1}$ | | Hilal et al. (2008) | Q | |
| | $5.9\times10^{-1}$ | | Modarresi et al. (2007) | Q | 67 |
| | | 7600 | Kühne et al. (2005) | Q | |
| | $7.0\times10^{-1}$ | | Yaffe et al. (2003) | Q | 248, 249 |
| | $4.7\times10^{-1}$ | | English and Carroll (2001) | Q | 230, 274 |
| | 1.9 | | Katritzky et al. (1998) | Q | |
| | $8.3\times10^{-1}$ | | Yaws et al. (1997) | Q | |
| | $6.0\times10^{-1}$ | | Suzuki et al. (1992) | Q | 232 |
| | $6.9\times10^{-1}$ | | Nirmalakhandan and Speece (1988) | Q | |
| | $7.0\times10^{-1}$ | | Duchowicz et al. (2020) | ? | 185, 21 |
| | | 6800 | Kühne et al. (2005) | ? | |
| | $8.4\times10^{-1}$ | | Yaws (1999) | ? | 21 |
| | $7.0\times10^{-1}$ | | Abraham et al. (1990) | ? | |
| (*S*)-2-methyl-1-butanol | $3.9\times10^{-1}$ | | Hilal et al. (2008) | Q | |
| $C_5H_{12}O$ | | | | | |
| [1565-80-6] | | | | | |
| QPRQEDXDYOZYLA-RXMQYKEDSA-N | | | | | |
| 3-methyl-1-butanol | $8.2\times10^{-1}$ | 8000 | Brockbank (2013) | L | 1 |
| $C_5H_{12}O$ | $7.0\times10^{-1}$ | | Plyasunov and Shock (2000) | L | |
| (isopentanol) | $6.7\times10^{-1}$ | 6900 | Ammari and Schroen (2019) | M | 11 |
| [123-51-3] | $8.2\times10^{-1}$ | 7700 | Fenclová et al. (2010) | M | 1 |
| PHTQWCKDNZKARW-UHFFFAOYSA-N | $3.3\times10^{-1}$ | | van Ruth et al. (2002) | M | 14 |
| | $3.2\times10^{-1}$ | | van Ruth and Villeneuve (2002) | M | 14, 361 |
| | $3.2\times10^{-1}$ | | van Ruth et al. (2001) | M | 14 |
| | $7.5\times10^{-1}$ | | Yaws (2003) | X | 258 |
| | $7.1\times10^{-1}$ | | Dupeux et al. (2022) | Q | 259 |
| | 1.6 | | Keshavarz et al. (2022) | Q | |
| | $1.2\times10^{-1}$ | | Abney (2021) | Q | 399 |
| | $5.3\times10^{-1}$ | | Duchowicz et al. (2020) | Q | 299 |



Table A3.2: Alcohols (ROH) (...continued)

| Substance Formula (Trivial Name) [CAS Registry Number] InChIKey | $H_s^{cp}$ (at $T^{\ominus}$) $\left[\dfrac{\mathrm{mol}}{\mathrm{m}^3\,\mathrm{Pa}}\right]$ | $\dfrac{\mathrm{d}\ln H_s^{cp}}{\mathrm{d}(1/T)}$ [K] | Reference | Type | Note |
|---|---|---|---|---|---|
| | $1.8\times10^{-1}$ | | Wang et al. (2017) | Q | 80, 238 |
| | $7.4\times10^{-1}$ | | Wang et al. (2017) | Q | 80, 239 |
| | $1.3$ | | Wang et al. (2017) | Q | 80, 240 |
| | $6.2\times10^{-1}$ | | Raventos-Duran et al. (2010) | Q | 271, 243 |
| | $4.9\times10^{-1}$ | | Raventos-Duran et al. (2010) | Q | 244 |
| | $7.8\times10^{-1}$ | | Raventos-Duran et al. (2010) | Q | 245 |
| | $4.6\times10^{-1}$ | | Hilal et al. (2008) | Q | |
| | $1.1$ | | Modarresi et al. (2007) | Q | 67 |
| | | 7600 | Kühne et al. (2005) | Q | |
| | $6.1\times10^{-1}$ | | Yaffe et al. (2003) | Q | 248, 272 |
| | $7.0\times10^{-1}$ | | Yao et al. (2002) | Q | 229 |
| | $4.7\times10^{-1}$ | | English and Carroll (2001) | Q | 230, 231 |
| | $6.9\times10^{-1}$ | | Nirmalakhandan et al. (1997) | Q | |
| | $7.4\times10^{-1}$ | | Yaws et al. (1997) | Q | |
| | $7.0\times10^{-1}$ | | Duchowicz et al. (2020) | ? | 185, 21 |
| | | 8200 | Kühne et al. (2005) | ? | |
| | $7.4\times10^{-1}$ | | Yaws (1999) | ? | 21 |
| | $7.0\times10^{-1}$ | | Abraham et al. (1990) | ? | |
| 2-methyl-2-butanol C$_5$H$_{12}$O (*tert*-pentanol) [75-85-4] MSXVEPNJUHWQHW-UHFFFAOYSA-N | $6.5\times10^{-1}$ | 7900 | Brockbank (2013) | L | 1 |
| | $7.1\times10^{-1}$ | 8200 | Plyasunov and Shock (2000) | L | |
| | $7.2\times10^{-1}$ | 8200 | Fenclová et al. (2010) | M | 1 |
| | $7.0\times10^{-1}$ | | Merk and Riederer (1997) | M | |
| | $7.1\times10^{-1}$ | | Butler et al. (1935) | M | |
| | $1.6$ | | Keshavarz et al. (2022) | Q | |
| | $2.3\times10^{-1}$ | | Duchowicz et al. (2020) | Q | |
| | $1.2\times10^{-1}$ | | Wang et al. (2017) | Q | 80, 238 |
| | $6.6\times10^{-1}$ | | Wang et al. (2017) | Q | 80, 239 |
| | $4.8\times10^{-1}$ | | Wang et al. (2017) | Q | 80, 240 |
| | $7.2\times10^{-1}$ | | HSDB (2015) | Q | 99 |
| | $6.2\times10^{-1}$ | | Raventos-Duran et al. (2010) | Q | 271, 243 |
| | $3.9\times10^{-1}$ | | Raventos-Duran et al. (2010) | Q | 244 |
| | $7.8\times10^{-1}$ | | Raventos-Duran et al. (2010) | Q | 245 |
| | $2.7\times10^{-1}$ | | Hilal et al. (2008) | Q | |
| | $1.1$ | | Modarresi et al. (2007) | Q | 67 |
| | | 7600 | Kühne et al. (2005) | Q | |
| | $7.9\times10^{-1}$ | | Yaffe et al. (2003) | Q | 248, 272 |
| | $2.8\times10^{-1}$ | | Yao et al. (2002) | Q | 229 |
| | $7.5\times10^{-1}$ | | English and Carroll (2001) | Q | 230, 231 |
| | $6.9\times10^{-1}$ | | Katritzky et al. (1998) | Q | |
| | $6.1\times10^{-1}$ | | Yaws et al. (1997) | Q | |
| | $5.0\times10^{-1}$ | | Suzuki et al. (1992) | Q | 232 |
| | $6.0\times10^{-1}$ | | Nirmalakhandan and Speece (1988) | Q | |
| | $7.2\times10^{-1}$ | | Duchowicz et al. (2020) | ? | 185, 21 |
| | | 7200 | Kühne et al. (2005) | ? | |
| | $5.6\times10^{-2}$ | | Yaws (1999) | ? | 21 |
| | $7.2\times10^{-1}$ | | Abraham et al. (1990) | ? | |





Table A3.2: Alcohols (ROH) (...continued)

| Substance Formula (Trivial Name) [CAS Registry Number] InChIKey | $H_s^{cp}$ (at $T^{\ominus}$) $\left[\dfrac{\text{mol}}{\text{m}^3\,\text{Pa}}\right]$ | $\dfrac{\text{d}\ln H_s^{cp}}{\text{d}(1/T)}$ [K] | Reference | Type | Note |
|---|---|---|---|---|---|
| 3-methyl-2-butanol | $5.1\times10^{-1}$ | 7600 | Brockbank (2013) | L | 1 |
| $C_5H_{12}O$ | $5.4\times10^{-1}$ | 7800 | Fenclová et al. (2010) | M | 1 |
| [598-75-4] | $5.6\times10^{-1}$ | | Duchowicz et al. (2020) | V | 186 |
| MXLMTQWGSQIYOW-UHFFFAOYSA-N | $2.1\times10^{-1}$ | | Duchowicz et al. (2020) | Q | |
| | $2.1\times10^{-1}$ | | Wang et al. (2017) | Q | 80, 238 |
| | $6.9\times10^{-1}$ | | Wang et al. (2017) | Q | 80, 239 |
| | $4.2\times10^{-1}$ | | Wang et al. (2017) | Q | 80, 240 |
| | $6.2\times10^{-1}$ | | Raventos-Duran et al. (2010) | Q | 271, 243 |
| | $3.9\times10^{-1}$ | | Raventos-Duran et al. (2010) | Q | 244 |
| | $7.8\times10^{-1}$ | | Raventos-Duran et al. (2010) | Q | 245 |
| | $3.1\times10^{-1}$ | | Hilal et al. (2008) | Q | |
| | $7.1\times10^{-1}$ | | Modarresi et al. (2007) | Q | 67 |
| | | 7600 | Kühne et al. (2005) | Q | |
| | $6.1\times10^{-1}$ | | Yaffe et al. (2003) | Q | 248, 249 |
| | $4.5\times10^{-1}$ | | Katritzky et al. (1998) | Q | |
| | $5.4\times10^{-1}$ | | Yaws et al. (1997) | Q | |
| | | 7500 | Kühne et al. (2005) | ? | |
| | $5.5\times10^{-1}$ | | Yaws (1999) | ? | 21 |
| 2,2-dimethyl-1-propanol | $3.0\times10^{-1}$ | 7800 | Brockbank (2013) | L | 1, 400 |
| $C_5H_{12}O$ | $1.9\times10^{-1}$ | | Duchowicz et al. (2020) | V | 186 |
| [75-84-3] | $1.9\times10^{-1}$ | | HSDB (2015) | V | |
| KPSSIOMAKSHJJG-UHFFFAOYSA-N | $5.7\times10^{-1}$ | | Yaws (2003) | X | 258 |
| | $3.7\times10^{-1}$ | | Dupeux et al. (2022) | Q | 259 |
| | $2.3\times10^{-1}$ | | Duchowicz et al. (2020) | Q | |
| | $1.2\times10^{-1}$ | | Wang et al. (2017) | Q | 80, 238 |
| | $5.9\times10^{-1}$ | | Wang et al. (2017) | Q | 80, 239 |
| | $3.0\times10^{-1}$ | | Wang et al. (2017) | Q | 80, 240 |
| | $6.2\times10^{-1}$ | | Raventos-Duran et al. (2010) | Q | 242, 243 |
| | $3.9\times10^{-1}$ | | Raventos-Duran et al. (2010) | Q | 244 |
| | $7.8\times10^{-1}$ | | Raventos-Duran et al. (2010) | Q | 245 |
| | $3.1\times10^{-1}$ | | Hilal et al. (2008) | Q | |
| | $5.7\times10^{-1}$ | | Modarresi et al. (2007) | Q | 67 |
| | | 7600 | Kühne et al. (2005) | Q | |
| | $4.9\times10^{-1}$ | | Saxena and Hildemann (1996) | E | 401 |
| | | 7900 | Kühne et al. (2005) | ? | |
| 1-hexanol | $7.6\times10^{-1}$ | 7800 | Brockbank (2013) | L | 1 |
| $C_6H_{14}O$ | $6.6\times10^{-1}$ | 8200 | Plyasunov and Shock (2000) | L | |
| [111-27-3] | $5.7\times10^{-1}$ | 7300 | Shunthirasingham et al. (2013) | M | |
| ZSIAUFGUXNUGDI-UHFFFAOYSA-N | $5.1\times10^{-1}$ | 6100 | Lei et al. (2007) | M | 395 |
| | $3.9\times10^{-1}$ | 5800 | Falabella et al. (2006) | M | 11, 338 |
| | $3.8\times10^{-1}$ | | Souchon et al. (2004) | M | |
| | $2.5\times10^{-1}$ | | van Ruth et al. (2002) | M | 14 |
| | $7.3\times10^{-1}$ | | van Ruth and Villeneuve (2002) | M | 14, 361 |
| | $1.6\times10^{-1}$ | | van Ruth et al. (2001) | M | 14 |
| | $3.9\times10^{-1}$ | 5800 | Gupta et al. (2000) | M | |
| | $9.8\times10^{-1}$ | | Altschuh et al. (1999) | M | |



Table A3.2: Alcohols (ROH) (...continued)

| Substance Formula (Trivial Name) [CAS Registry Number] InChIKey | $H_s^{cp}$ (at $T^\ominus$) $\left[\dfrac{\mathrm{mol}}{\mathrm{m^3\,Pa}}\right]$ | $\dfrac{\mathrm{d}\ln H_s^{cp}}{\mathrm{d}(1/T)}$ [K] | Reference | Type | Note |
|---|---|---|---|---|---|
| | $3.5\times10^{-1}$ | | Eger et al. (1999) | M | 14 |
| | $5.8\times10^{-1}$ | | Merk and Riederer (1997) | M | |
| | $6.4\times10^{-1}$ | | Li and Carr (1993) | M | |
| | $6.9\times10^{-1}$ | | Rytting et al. (1978) | M | |
| | $5.8\times10^{-1}$ | | Buttery et al. (1969) | M | |
| | $5.3\times10^{-1}$ | | Mackay et al. (2006c) | V | |
| | $5.3\times10^{-1}$ | | Mackay et al. (1995) | V | |
| | $7.6\times10^{-1}$ | | Hwang et al. (1992) | V | |
| | 1.7 | 5100 | Djerki and Laub (1988) | V | |
| | | 8200 | Abraham (1984) | V | |
| | $6.4\times10^{-1}$ | | Hine and Mookerjee (1975) | V | |
| | $6.4\times10^{-1}$ | | Butler et al. (1935) | V | |
| | $5.2\times10^{-1}$ | | Yaws (2003) | X | 258 |
| | $5.8\times10^{-1}$ | | Dupeux et al. (2022) | Q | 259 |
| | $2.1\times10^{-1}$ | | Keshavarz et al. (2022) | Q | |
| | 1.4 | | Duchowicz et al. (2020) | Q | 299 |
| | $1.4\times10^{-1}$ | | Wang et al. (2017) | Q | 80, 238 |
| | $5.0\times10^{-1}$ | | Wang et al. (2017) | Q | 80, 239 |
| | $8.5\times10^{-1}$ | | Wang et al. (2017) | Q | 80, 240 |
| | $6.4\times10^{-1}$ | | Li et al. (2014) | Q | 241 |
| | $4.9\times10^{-1}$ | | Raventos-Duran et al. (2010) | Q | 271, 243 |
| | $3.1\times10^{-1}$ | | Raventos-Duran et al. (2010) | Q | 244 |
| | $6.2\times10^{-1}$ | | Raventos-Duran et al. (2010) | Q | 245 |
| | $3.7\times10^{-1}$ | | Hilal et al. (2008) | Q | |
| | 1.1 | | Modarresi et al. (2007) | Q | 67 |
| | | 7900 | Kühne et al. (2005) | Q | |
| | $4.1\times10^{-1}$ | | Yao et al. (2002) | Q | 229, 267 |
| | $6.2\times10^{-1}$ | | English and Carroll (2001) | Q | 230, 231 |
| | 1.5 | | Katritzky et al. (1998) | Q | |
| | $4.7\times10^{-1}$ | | Yaws et al. (1997) | Q | |
| | $3.9\times10^{-1}$ | | Russell et al. (1992) | Q | 279 |
| | $5.1\times10^{-1}$ | | Suzuki et al. (1992) | Q | 232 |
| | $6.2\times10^{-1}$ | | Nirmalakhandan and Speece (1988) | Q | |
| | $5.8\times10^{-1}$ | | Duchowicz et al. (2020) | ? | 185, 21 |
| | | 8400 | Kühne et al. (2005) | ? | |
| | $4.7\times10^{-1}$ | | Yaws (1999) | ? | 21 |
| | $5.3\times10^{-1}$ | | Yaws and Yang (1992) | ? | 21 |
| | $6.9\times10^{-1}$ | | Abraham et al. (1990) | ? | |
| 2-hexanol C$_6$H$_{14}$O [626-93-7] QNVRIHYSUZMSGM-UHFFFAOYSA-N | $3.9\times10^{-1}$ | 8100 | Brockbank (2013) | L | 1 |
| | $5.2\times10^{-1}$ | | Plyasunov and Shock (2000) | L | |
| | $4.8\times10^{-1}$ | | Merk and Riederer (1997) | M | |
| | $4.0\times10^{-1}$ | | Duchowicz et al. (2020) | V | 186 |
| | $4.2\times10^{-1}$ | | Yaws (2003) | X | 258 |
| | $7.3\times10^{-1}$ | | Dupeux et al. (2022) | Q | 259 |
| | $5.4\times10^{-1}$ | | Duchowicz et al. (2020) | Q | |
| | $1.7\times10^{-1}$ | | Wang et al. (2017) | Q | 80, 238 |
| | $4.6\times10^{-1}$ | | Wang et al. (2017) | Q | 80, 239 |



Table A3.2: Alcohols (ROH) (...continued)

| Substance Formula (Trivial Name) [CAS Registry Number] InChIKey | $H_s^{cp}$ (at $T^\ominus$) $\left[\dfrac{\text{mol}}{\text{m}^3\,\text{Pa}}\right]$ | $\dfrac{\text{d}\ln H_s^{cp}}{\text{d}(1/T)}$ [K] | Reference | Type | Note |
|---|---|---|---|---|---|
| | $5.6\times10^{-1}$ | | Wang et al. (2017) | Q | 80, 240 |
| | $4.9\times10^{-1}$ | | Raventos-Duran et al. (2010) | Q | 271, 243 |
| | $2.5\times10^{-1}$ | | Raventos-Duran et al. (2010) | Q | 244 |
| | $6.2\times10^{-1}$ | | Raventos-Duran et al. (2010) | Q | 245 |
| | $2.5\times10^{-1}$ | | Hilal et al. (2008) | Q | |
| | $2.8\times10^{-1}$ | | Modarresi et al. (2007) | Q | 67 |
| | $4.0\times10^{-1}$ | | Yaffe et al. (2003) | Q | 248, 249 |
| | $2.8\times10^{-1}$ | | Yao et al. (2002) | Q | 229 |
| | $1.1$ | | Katritzky et al. (1998) | Q | |
| | $4.2\times10^{-1}$ | | Yaws et al. (1997) | Q | |
| | $4.2\times10^{-1}$ | | Yaws (1999) | ? | 21 |
| 3-hexanol $C_6H_{14}O$ [623-37-0] ZOCHHNOQQHDWHG-UHFFFAOYSA-N | $3.7\times10^{-1}$ | 8400 | Brockbank (2013) | L | 1 |
| | $4.2\times10^{-1}$ | 8400 | Plyasunov and Shock (2000) | L | |
| | $2.5\times10^{-1}$ | | Duchowicz et al. (2020) | V | 186 |
| | $2.3\times10^{-1}$ | | Meylan and Howard (1991) | V | |
| | $2.0\times10^{-1}$ | | Hine and Mookerjee (1975) | V | |
| | $3.9\times10^{-1}$ | 8400 | Cabani et al. (1975b) | T | |
| | $2.2\times10^{-1}$ | | Yaws (2003) | X | 237 |
| | $5.4\times10^{-1}$ | | Duchowicz et al. (2020) | Q | |
| | $1.7\times10^{-1}$ | | Wang et al. (2017) | Q | 80, 238 |
| | $5.3\times10^{-1}$ | | Wang et al. (2017) | Q | 80, 239 |
| | $2.5\times10^{-1}$ | | Wang et al. (2017) | Q | 80, 240 |
| | $4.9\times10^{-1}$ | | Raventos-Duran et al. (2010) | Q | 242, 243 |
| | $3.1\times10^{-1}$ | | Raventos-Duran et al. (2010) | Q | 244 |
| | $6.2\times10^{-1}$ | | Raventos-Duran et al. (2010) | Q | 245 |
| | $2.0\times10^{-1}$ | | Gharagheizi et al. (2010) | Q | 246 |
| | $2.8\times10^{-1}$ | | Hilal et al. (2008) | Q | |
| | $4.1\times10^{-1}$ | | Modarresi et al. (2007) | Q | 67 |
| | $4.0\times10^{-1}$ | | Yaffe et al. (2003) | Q | 248, 249 |
| | $5.8\times10^{-1}$ | | English and Carroll (2001) | Q | 230, 231 |
| | $4.1\times10^{-1}$ | | Yaws et al. (1997) | Q | |
| | $4.8\times10^{-1}$ | | Suzuki et al. (1992) | Q | 232 |
| | $5.6\times10^{-1}$ | | Meylan and Howard (1991) | Q | |
| | $6.0\times10^{-1}$ | | Nirmalakhandan and Speece (1988) | Q | |
| | $3.9\times10^{-1}$ | | Abraham et al. (1990) | ? | |
| 2-methyl-1-pentanol $C_6H_{14}O$ [105-30-6] PFNHSEQQEPMLNI-UHFFFAOYSA-N | $4.6\times10^{-1}$ | | Plyasunov and Shock (2000) | L | |
| | $2.3\times10^{-1}$ | | Duchowicz et al. (2020) | V | 186 |
| | $2.3\times10^{-1}$ | | HSDB (2015) | V | |
| | $5.4\times10^{-1}$ | | Duchowicz et al. (2020) | Q | |
| | $1.7\times10^{-1}$ | | Wang et al. (2017) | Q | 80, 238 |
| | $6.0\times10^{-1}$ | | Wang et al. (2017) | Q | 80, 239 |
| | $5.5\times10^{-1}$ | | Wang et al. (2017) | Q | 80, 240 |
| | $4.9\times10^{-1}$ | | Raventos-Duran et al. (2010) | Q | 271, 243 |
| | $3.9\times10^{-1}$ | | Raventos-Duran et al. (2010) | Q | 244 |
| | $6.2\times10^{-1}$ | | Raventos-Duran et al. (2010) | Q | 245 |
| | $4.4\times10^{-1}$ | | Hilal et al. (2008) | Q | |



Table A3.2: Alcohols (ROH) (...continued)

| Substance<br>Formula<br>(Trivial Name)<br>[CAS Registry Number]<br>InChIKey | $H_s^{cp}$<br>(at $T^{\ominus}$)<br>$\left[\dfrac{\text{mol}}{\text{m}^3\,\text{Pa}}\right]$ | $\dfrac{\text{d}\ln H_s^{cp}}{\text{d}(1/T)}$<br><br>[K] | Reference | Type | Note |
|---|---|---|---|---|---|
| | $4.6\times10^{-1}$ | | Modarresi et al. (2007) | Q | 67 |
| | $3.1\times10^{-1}$ | | Yaws et al. (1997) | Q | |
| | $3.1\times10^{-1}$ | | Yaws (1999) | ? | 21 |
| 3-methyl-1-pentanol<br>$C_6H_{14}O$<br>[589-35-5]<br>IWTBVKIGCDZRPL-UHFFFAOYSA-N | $7.5\times10^{-1}$<br>$1.7\times10^{-1}$<br>$6.8\times10^{-1}$<br>$9.6\times10^{-1}$<br>$3.3\times10^{-1}$<br>$3.8\times10^{-1}$ | 8600 | Brockbank (2013)<br>Wang et al. (2017)<br>Wang et al. (2017)<br>Wang et al. (2017)<br>Gharagheizi et al. (2012)<br>Hilal et al. (2008) | L<br>Q<br>Q<br>Q<br>Q<br>Q | 1<br>80, 238<br>80, 239<br>80, 240<br><br> |
| 2-methyl-2-pentanol<br>$C_6H_{14}O$<br>[590-36-3]<br>WFRBDWRZVBPBDO-UHFFFAOYSA-N | $2.8\times10^{-1}$<br>$3.1\times10^{-1}$<br>$2.4\times10^{-1}$<br>$1.0\times10^{-1}$<br>$5.1\times10^{-1}$<br>$3.9\times10^{-1}$<br>$4.9\times10^{-1}$<br>$3.1\times10^{-1}$<br>$6.2\times10^{-1}$<br>$3.2\times10^{-1}$<br>$7.7\times10^{-1}$<br>$2.9\times10^{-1}$<br>$6.2\times10^{-1}$<br>$6.7\times10^{-1}$<br>$5.0\times10^{-1}$<br>$3.9\times10^{-1}$<br>$4.7\times10^{-1}$<br>$3.1\times10^{-1}$ | | Duchowicz et al. (2020)<br>Hine and Mookerjee (1975)<br>Duchowicz et al. (2020)<br>Wang et al. (2017)<br>Wang et al. (2017)<br>Wang et al. (2017)<br>Raventos-Duran et al. (2010)<br>Raventos-Duran et al. (2010)<br>Raventos-Duran et al. (2010)<br>Hilal et al. (2008)<br>Modarresi et al. (2007)<br>Yaffe et al. (2003)<br>English and Carroll (2001)<br>Katritzky et al. (1998)<br>Yaws et al. (1997)<br>Suzuki et al. (1992)<br>Nirmalakhandan and Speece (1988)<br>Abraham et al. (1990) | V<br>V<br>Q<br>Q<br>Q<br>Q<br>Q<br>Q<br>Q<br>Q<br>Q<br>Q<br>Q<br>Q<br>Q<br>Q<br>Q<br>? | 186<br><br><br>80, 238<br>80, 239<br>80, 240<br>242, 243<br>244<br>245<br><br>67<br>248, 249<br>230, 231<br><br><br>232<br><br> |
| 3-methyl-2-pentanol<br>$C_6H_{14}O$<br>[565-60-6]<br>ZXNBBWHRUSXUFZ-UHFFFAOYSA-N | $5.0\times10^{-1}$<br>$1.7\times10^{-1}$<br>$8.1\times10^{-1}$<br>$3.6\times10^{-1}$<br>$4.0\times10^{-1}$<br>$2.8\times10^{-1}$ | | Plyasunov and Shock (2000)<br>Wang et al. (2017)<br>Wang et al. (2017)<br>Wang et al. (2017)<br>Yaffe et al. (2003)<br>Yaws et al. (1997) | L<br>Q<br>Q<br>Q<br>Q<br>Q | <br>80, 238<br>80, 239<br>80, 240<br>248, 272<br> |
| 4-methyl-2-pentanol<br>$C_6H_{14}O$<br>[108-11-2]<br>WVYWICLMDOOCFB-UHFFFAOYSA-N | $2.2\times10^{-1}$<br>$3.5\times10^{-1}$<br>$2.1\times10^{-1}$<br>$2.2\times10^{-1}$<br>$2.1\times10^{-1}$<br>$2.1\times10^{-1}$<br>$1.7\times10^{-1}$<br>$5.6\times10^{-1}$<br>$4.7\times10^{-1}$<br>$4.9\times10^{-1}$<br>$3.1\times10^{-1}$<br>$6.2\times10^{-1}$ | 7300 | Brockbank (2013)<br>Plyasunov and Shock (2000)<br>Meylan and Howard (1991)<br>Hine and Mookerjee (1975)<br>Keshavarz et al. (2022)<br>Duchowicz et al. (2020)<br>Wang et al. (2017)<br>Wang et al. (2017)<br>Wang et al. (2017)<br>Raventos-Duran et al. (2010)<br>Raventos-Duran et al. (2010)<br>Raventos-Duran et al. (2010) | L<br>L<br>V<br>V<br>Q<br>Q<br>Q<br>Q<br>Q<br>Q<br>Q<br>Q | 1<br><br><br><br><br><br>80, 238<br>80, 239<br>80, 240<br>242, 243<br>244<br>245 |





Table A3.2: Alcohols (ROH) (... continued)

| Substance Formula (Trivial Name) [CAS Registry Number] InChIKey | $H_s^{cp}$ (at $T^{\ominus}$) $\left[\dfrac{\mathrm{mol}}{\mathrm{m^3\,Pa}}\right]$ | $\dfrac{\mathrm{d}\ln H_s^{cp}}{\mathrm{d}(1/T)}$ [K] | Reference | Type | Note |
|---|---|---|---|---|---|
| | $2.6\times10^{-1}$ | | Hilal et al. (2008) | Q | |
| | $2.8\times10^{-1}$ | | Modarresi et al. (2007) | Q | 67 |
| | | 7900 | Kühne et al. (2005) | Q | |
| | $2.3\times10^{-1}$ | | Yaffe et al. (2003) | Q | 248, 249 |
| | $5.3\times10^{-1}$ | | Katritzky et al. (1998) | Q | |
| | $1.9\times10^{-1}$ | | Yaws et al. (1997) | Q | |
| | $4.0\times10^{-1}$ | | Suzuki et al. (1992) | Q | 232 |
| | $5.6\times10^{-1}$ | | Meylan and Howard (1991) | Q | |
| | $4.8\times10^{-1}$ | | Nirmalakhandan and Speece (1988) | Q | |
| | $2.2\times10^{-1}$ | | Duchowicz et al. (2020) | ? | 185, 21 |
| | | 8700 | Kühne et al. (2005) | ? | |
| | $2.1\times10^{-1}$ | | Yaws (1999) | ? | 21, 402 |
| | $2.2\times10^{-1}$ | | Abraham et al. (1990) | ? | |
| 2-methyl-3-pentanol $C_6H_{14}O$ [565-67-3] ISTJMQSHILQAEC-UHFFFAOYSA-N | $2.9\times10^{-1}$ | | Hine and Mookerjee (1975) | V | |
| | $1.7\times10^{-1}$ | | Wang et al. (2017) | Q | 80, 238 |
| | $6.9\times10^{-1}$ | | Wang et al. (2017) | Q | 80, 239 |
| | $1.4\times10^{-1}$ | | Wang et al. (2017) | Q | 80, 240 |
| | $3.3\times10^{-1}$ | | Hilal et al. (2008) | Q | |
| | $5.8\times10^{-1}$ | | Modarresi et al. (2007) | Q | 67 |
| | $2.9\times10^{-1}$ | | Yaffe et al. (2003) | Q | 248, 249 |
| | $3.7\times10^{-1}$ | | English and Carroll (2001) | Q | 230, 274 |
| | $3.7\times10^{-1}$ | | Yaws et al. (1997) | Q | |
| | $4.2\times10^{-1}$ | | Suzuki et al. (1992) | Q | 232 |
| | $5.2\times10^{-1}$ | | Nirmalakhandan and Speece (1988) | Q | |
| | $2.9\times10^{-1}$ | | Abraham et al. (1990) | ? | |
| 3-methyl-3-pentanol $C_6H_{14}O$ [77-74-7] FRDAATYAJDYRNW-UHFFFAOYSA-N | $2.4\times10^{-1}$ | 7300 | Brockbank (2013) | L | 1 |
| | $4.7\times10^{-1}$ | | Plyasunov and Shock (2000) | L | |
| | $4.8\times10^{-1}$ | | Merk and Riederer (1997) | M | |
| | $5.6\times10^{-1}$ | | Duchowicz et al. (2020) | V | 186 |
| | $2.4\times10^{-1}$ | | Duchowicz et al. (2020) | Q | |
| | $1.0\times10^{-1}$ | | Wang et al. (2017) | Q | 80, 238 |
| | $5.5\times10^{-1}$ | | Wang et al. (2017) | Q | 80, 239 |
| | $2.8\times10^{-1}$ | | Wang et al. (2017) | Q | 80, 240 |
| | $4.9\times10^{-1}$ | | Raventos-Duran et al. (2010) | Q | 242, 243 |
| | $3.1\times10^{-1}$ | | Raventos-Duran et al. (2010) | Q | 244 |
| | $6.2\times10^{-1}$ | | Raventos-Duran et al. (2010) | Q | 245 |
| | $2.1\times10^{-1}$ | | Hilal et al. (2008) | Q | |
| | $3.3\times10^{-1}$ | | Modarresi et al. (2007) | Q | 67 |
| | $2.9\times10^{-1}$ | | Yaffe et al. (2003) | Q | 248, 272 |
| | $5.6\times10^{-1}$ | | Katritzky et al. (1998) | Q | |
| | $7.0\times10^{-1}$ | | Yaws et al. (1997) | Q | |





Table A3.2: Alcohols (ROH) (. . . continued)

| Substance<br>Formula<br>(Trivial Name)<br>[CAS Registry Number]<br>InChIKey | $H_s^{cp}$<br>(at $T^\ominus$)<br>$\left[\dfrac{\text{mol}}{\text{m}^3\,\text{Pa}}\right]$ | $\dfrac{\text{d}\ln H_s^{cp}}{\text{d}(1/T)}$<br><br>[K] | Reference | Type | Note |
|---|---|---|---|---|---|
| 2-ethyl-1-butanol | $5.4\times10^{-1}$ | | Plyasunov and Shock (2000) | L | |
| $C_6H_{14}O$ | $1.9\times10^{-1}$ | | Duchowicz et al. (2020) | V | 186 |
| [97-95-0] | $5.4\times10^{-1}$ | | Duchowicz et al. (2020) | Q | |
| TZYRSLHNPKPEFV-UHFFFAOYSA-N | $4.9\times10^{-1}$ | | Raventos-Duran et al. (2010) | Q | 242, 243 |
| | $3.9\times10^{-1}$ | | Raventos-Duran et al. (2010) | Q | 244 |
| | $6.2\times10^{-1}$ | | Raventos-Duran et al. (2010) | Q | 245 |
| | $4.7\times10^{-1}$ | | Hilal et al. (2008) | Q | |
| | $5.5\times10^{-1}$ | | Modarresi et al. (2007) | Q | 67 |
| | $3.5\times10^{-1}$ | | Yao et al. (2002) | Q | 229 |
| | 2.0 | | Katritzky et al. (1998) | Q | |
| | $4.8\times10^{-1}$ | | Yaws et al. (1997) | Q | |
| | $4.8\times10^{-1}$ | | Yaws (1999) | ? | 21 |
| 2,2-dimethyl-1-butanol | $1.0\times10^{-1}$ | | Wang et al. (2017) | Q | 80, 238 |
| $C_6H_{14}O$ | $5.4\times10^{-1}$ | | Wang et al. (2017) | Q | 80, 239 |
| [1185-33-7] | $2.7\times10^{-1}$ | | Wang et al. (2017) | Q | 80, 240 |
| XRMVWAKMXZNZIL-UHFFFAOYSA-N | $4.0\times10^{-1}$ | | Yaffe et al. (2003) | Q | 248, 272 |
| | $2.8\times10^{-1}$ | | Yaws et al. (1997) | Q | |
| 2,3-dimethyl-1-butanol | $1.7\times10^{-1}$ | | Wang et al. (2017) | Q | 80, 238 |
| $C_6H_{14}O$ | $9.1\times10^{-1}$ | | Wang et al. (2017) | Q | 80, 239 |
| [19550-30-2] | $5.1\times10^{-1}$ | | Wang et al. (2017) | Q | 80, 240 |
| SXSWMAUXEHKFGX-UHFFFAOYSA-N | $8.2\times10^{-1}$ | | Yaws et al. (1997) | Q | |
| | $4.2\times10^{-1}$ | | Suzuki et al. (1992) | Q | 403, 232 |
| | $4.7\times10^{-1}$ | | Nirmalakhandan and Speece (1988) | Q | |
| 2,3-dimethyl-2-butanol | $3.8\times10^{-1}$ | | Plyasunov and Shock (2000) | L | |
| $C_6H_{14}O$ | $9.9\times10^{-1}$ | | Duchowicz et al. (2020) | V | 186 |
| [594-60-5] | $3.0\times10^{-1}$ | | Hine and Mookerjee (1975) | V | 404 |
| IKECULIHBUCAKR-UHFFFAOYSA-N | $9.3\times10^{-2}$ | | Duchowicz et al. (2020) | Q | |
| | $1.2\times10^{-1}$ | | Wang et al. (2017) | Q | 80, 238 |
| | $6.3\times10^{-1}$ | | Wang et al. (2017) | Q | 80, 239 |
| | $3.0\times10^{-1}$ | | Wang et al. (2017) | Q | 80, 240 |
| | $2.0\times10^{-1}$ | | Hilal et al. (2008) | Q | |
| | $2.4\times10^{-1}$ | | Modarresi et al. (2007) | Q | 67 |
| | 1.0 | | Yaffe et al. (2003) | Q | 248, 249 |
| | $6.0\times10^{-1}$ | | Katritzky et al. (1998) | Q | |
| 3,3-dimethyl-1-butanol | $1.0\times10^{-1}$ | | Wang et al. (2017) | Q | 80, 238 |
| $C_6H_{14}O$ | $5.6\times10^{-1}$ | | Wang et al. (2017) | Q | 80, 239 |
| [624-95-3] | 1.6 | | Wang et al. (2017) | Q | 80, 240 |
| DUXCSEISVMREAX-UHFFFAOYSA-N | $2.4\times10^{-1}$ | | Gharagheizi et al. (2012) | Q | |
| 3,3-dimethyl-2-butanol | $1.2\times10^{-1}$ | | Wang et al. (2017) | Q | 80, 238 |
| $C_6H_{14}O$ | $5.3\times10^{-1}$ | | Wang et al. (2017) | Q | 80, 239 |
| [464-07-3] | $2.0\times10^{-1}$ | | Wang et al. (2017) | Q | 80, 240 |
| DFOXKPDFWGNLJU-UHFFFAOYSA-N | $5.6\times10^{-1}$ | | HSDB (2015) | Q | 99 |
| | $4.0\times10^{-1}$ | | Yaffe et al. (2003) | Q | 248, 249 |
| | $4.9\times10^{-1}$ | | Yaws et al. (1997) | Q | |



Table A3.2: Alcohols (ROH) (...continued)

| Substance Formula (Trivial Name) [CAS Registry Number] InChIKey | $H_s^{cp}$ (at $T^{\ominus}$) $\left[\dfrac{\mathrm{mol}}{\mathrm{m^3\,Pa}}\right]$ | $\dfrac{\mathrm{d}\ln H_s^{cp}}{\mathrm{d}(1/T)}$ [K] | Reference | Type | Note |
|---|---|---|---|---|---|
| 2,2-bis(hydroxymethyl)-1-butanol | $1.2\times10^6$ | | Duchowicz et al. (2020) | V | 186 |
| $C_6H_{14}O_3$ | $8.8\times10^5$ | | Duchowicz et al. (2020) | Q | |
| [77-99-6] | | | | | |
| ZJCCRDAZUWHFQH-UHFFFAOYSA-N | | | | | |
| 1-heptanol | $5.3\times10^{-1}$ | 8200 | Brockbank (2013) | L | 1 |
| $C_7H_{16}O$ | $5.4\times10^{-1}$ | 8600 | Plyasunov and Shock (2000) | L | |
| [111-70-6] | $3.8\times10^{-1}$ | 7200 | Shunthirasingham et al. (2013) | M | |
| BBMCTIGTTCKYKF-UHFFFAOYSA-N | $3.6\times10^{-1}$ | 6300 | Lei et al. (2007) | M | 395 |
| | $8.6\times10^{-1}$ | | Altschuh et al. (1999) | M | |
| | $1.8\times10^{-1}$ | | Shiu and Mackay (1997) | M | |
| | $6.2\times10^{-1}$ | | Mackay et al. (2006c) | V | |
| | $6.2\times10^{-1}$ | | Shiu and Mackay (1997) | V | |
| | $6.2\times10^{-1}$ | | Mackay et al. (1995) | V | |
| | 4.6 | 5300 | Djerki and Laub (1988) | V | |
| | $4.9\times10^{-1}$ | 8700 | Abraham (1984) | V | |
| | $5.3\times10^{-1}$ | | Hine and Mookerjee (1975) | V | |
| | $5.2\times10^{-1}$ | | Butler et al. (1935) | V | |
| | $5.2\times10^{-1}$ | | Yaws (2003) | X | 258 |
| | $4.6\times10^{-1}$ | | Dupeux et al. (2022) | Q | 259 |
| | $2.8\times10^{-1}$ | | Keshavarz et al. (2022) | Q | |
| | 1.4 | | Duchowicz et al. (2020) | Q | |
| | $3.9\times10^{-1}$ | | Raventos-Duran et al. (2010) | Q | 271, 243 |
| | $2.5\times10^{-1}$ | | Raventos-Duran et al. (2010) | Q | 244 |
| | $3.9\times10^{-1}$ | | Raventos-Duran et al. (2010) | Q | 245 |
| | $3.0\times10^{-1}$ | | Hilal et al. (2008) | Q | |
| | $7.3\times10^{-1}$ | | Modarresi et al. (2007) | Q | 67 |
| | | 8300 | Kühne et al. (2005) | Q | |
| | $1.9\times10^{-1}$ | | Yaffe et al. (2003) | Q | 248, 249 |
| | $2.9\times10^{-1}$ | | Yao et al. (2002) | Q | 229 |
| | $5.0\times10^{-1}$ | | English and Carroll (2001) | Q | 230, 231 |
| | 1.3 | | Katritzky et al. (1998) | Q | |
| | $5.2\times10^{-1}$ | | Yaws et al. (1997) | Q | |
| | $1.8\times10^{-1}$ | | Russell et al. (1992) | Q | 279 |
| | $3.9\times10^{-1}$ | | Suzuki et al. (1992) | Q | 232 |
| | $5.0\times10^{-1}$ | | Nirmalakhandan and Speece (1988) | Q | |
| | $3.8\times10^{-1}$ | | Rumble (2021) | ? | 405 |
| | $5.2\times10^{-1}$ | | Duchowicz et al. (2020) | ? | 185, 21 |
| | | 9400 | Kühne et al. (2005) | ? | |
| | $5.2\times10^{-1}$ | | Yaws (1999) | ? | 21 |
| | $8.5\times10^{-1}$ | | Yaws and Yang (1992) | ? | 21 |
| | $5.0\times10^{-1}$ | | Abraham et al. (1990) | ? | |





Table A3.2: Alcohols (ROH) (...continued)

| Substance Formula (Trivial Name) [CAS Registry Number] InChIKey | $H_s^{cp}$ (at $T^\ominus$) $\left[\dfrac{\text{mol}}{\text{m}^3\,\text{Pa}}\right]$ | $\dfrac{\text{d}\ln H_s^{cp}}{\text{d}(1/T)}$ [K] | Reference | Type | Note |
|---|---|---|---|---|---|
| 2-heptanol | $3.3\times10^{-1}$ | 8700 | Brockbank (2013) | L | 1 |
| $C_7H_{16}O$ | $4.1\times10^{-1}$ | | Plyasunov and Shock (2000) | L | |
| [543-49-7] | $1.8\times10^{-1}$ | | Duchowicz et al. (2020) | V | 186 |
| CETWDUZRCINIHU-UHFFFAOYSA-N | $1.2\times10^{-1}$ | | Yaws (2003) | X | 258 |
| | $1.2\times10^{-1}$ | | Yaws (2003) | X | 237, 38 |
| | $5.8\times10^{-1}$ | | Dupeux et al. (2022) | Q | 259 |
| | $5.5\times10^{-1}$ | | Duchowicz et al. (2020) | Q | |
| | $3.9\times10^{-1}$ | | Raventos-Duran et al. (2010) | Q | 242, 243 |
| | $2.0\times10^{-1}$ | | Raventos-Duran et al. (2010) | Q | 244 |
| | $3.9\times10^{-1}$ | | Raventos-Duran et al. (2010) | Q | 245 |
| | $1.2\times10^{-1}$ | | Gharagheizi et al. (2010) | Q | 246 |
| | $2.0\times10^{-1}$ | | Hilal et al. (2008) | Q | |
| | $2.5\times10^{-1}$ | | Modarresi et al. (2007) | Q | 67 |
| | $2.6\times10^{-1}$ | | Yao et al. (2002) | Q | 229 |
| | $1.2\times10^{-1}$ | | Yaws et al. (1997) | Q | |
| | $1.7\times10^{-1}$ | | Yaws (1999) | ? | 21, 38 |
| 3-heptanol | $3.2\times10^{-1}$ | 9100 | Brockbank (2013) | L | 1 |
| $C_7H_{16}O$ | $3.1\times10^{-1}$ | | Plyasunov and Shock (2000) | L | |
| [589-82-2] | $3.5\times10^{-1}$ | | Duchowicz et al. (2020) | V | 186 |
| RZKSECIXORKHQS-UHFFFAOYSA-N | $5.5\times10^{-1}$ | | Duchowicz et al. (2020) | Q | |
| | $1.3\times10^{-1}$ | | Wang et al. (2017) | Q | 80, 238 |
| | $4.1\times10^{-1}$ | | Wang et al. (2017) | Q | 80, 239 |
| | $2.2\times10^{-1}$ | | Wang et al. (2017) | Q | 80, 240 |
| | $1.9\times10^{-1}$ | | Gharagheizi et al. (2012) | Q | |
| | $3.9\times10^{-1}$ | | Raventos-Duran et al. (2010) | Q | 242, 243 |
| | $2.5\times10^{-1}$ | | Raventos-Duran et al. (2010) | Q | 244 |
| | $3.9\times10^{-1}$ | | Raventos-Duran et al. (2010) | Q | 245 |
| | $4.0\times10^{-1}$ | | Modarresi et al. (2007) | Q | 67 |
| | $3.7\times10^{-1}$ | | Yaffe et al. (2003) | Q | 248, 249 |
| | $5.3\times10^{-1}$ | | Katritzky et al. (1998) | Q | |
| | $2.1\times10^{-1}$ | | Yaws et al. (1997) | Q | |
| 4-heptanol | $3.4\times10^{-1}$ | 9100 | Plyasunov and Shock (2000) | L | |
| $C_7H_{16}O$ | $3.5\times10^{-1}$ | 9100 | Cabani et al. (1975b) | T | |
| [589-55-9] | $1.8\times10^{-1}$ | | Gharagheizi et al. (2012) | Q | |
| YVBCULSIZWMTFY-UHFFFAOYSA-N | $2.2\times10^{-1}$ | | Yaws et al. (1997) | Q | |
| 2-methyl-1-hexanol | $6.9\times10^{-1}$ | 11000 | Hiatt (2013) | M | |
| $C_7H_{16}O$ | $1.7\times10^{-1}$ | | Yaws et al. (1997) | Q | |
| [624-22-6] | | | | | |
| LCFKURIJYIJNRU-UHFFFAOYSA-N | | | | | |
| 3-methyl-1-hexanol | $1.3\times10^{-1}$ | | Yaws et al. (1997) | Q | |
| $C_7H_{16}O$ | | | | | |
| [13231-81-7] | | | | | |
| YGZVAQICDGBHMD-UHFFFAOYSA-N | | | | | |



Table A3.2: Alcohols (ROH) (...continued)

| Substance Formula (Trivial Name) [CAS Registry Number] InChIKey | $H_s^{cp}$ (at $T^{\ominus}$) $\left[\dfrac{\text{mol}}{\text{m}^3\,\text{Pa}}\right]$ | $\dfrac{\text{d}\ln H_s^{cp}}{\text{d}(1/T)}$ [K] | Reference | Type | Note |
|---|---|---|---|---|---|
| 4-methyl-1-hexanol $C_7H_{16}O$ [818-49-5] YNPVNLWKVZZBTM-UHFFFAOYSA-N | $1.3\times10^{-1}$ | | Yaws et al. (1997) | Q | |
| 5-methyl-1-hexanol $C_7H_{16}O$ [627-98-5] ZVHAANQOQZVVFD-UHFFFAOYSA-N | $2.8\times10^{-1}$ | | Yaws et al. (1997) | Q | |
| 2-methyl-2-hexanol $C_7H_{16}O$ [625-23-0] KRIMXCDMVRMCTC-UHFFFAOYSA-N | $9.1\times10^{-2}$ $4.0\times10^{-1}$ $3.8\times10^{-1}$ $6.7\times10^{-1}$ $6.4\times10^{-1}$ | | Wang et al. (2017) Wang et al. (2017) Wang et al. (2017) Yaffe et al. (2003) Yaws et al. (1997) | Q Q Q Q Q | 80, 238 80, 239 80, 240 248, 249 |
| 3-methyl-2-hexanol $C_7H_{16}O$ [2313-65-7] IRLSKJITMWPWNY-UHFFFAOYSA-N | $1.5\times10^{-1}$ $1.7\times10^{-1}$ $1.2\times10^{-1}$ $4.9\times10^{-1}$ | | Yaws (2003) Gharagheizi et al. (2012) Gharagheizi et al. (2010) Yaws et al. (1997) | X Q Q Q | 237 246 |
| 4-methyl-2-hexanol $C_7H_{16}O$ [2313-61-3] KZUBXUKRWLMPIO-UHFFFAOYSA-N | $1.6\times10^{-1}$ $5.1\times10^{-1}$ $4.7\times10^{-1}$ $5.0\times10^{-1}$ | | Wang et al. (2017) Wang et al. (2017) Wang et al. (2017) Yaws et al. (1997) | Q Q Q Q | 80, 238 80, 239 80, 240 |
| 5-methyl-2-hexanol $C_7H_{16}O$ [627-59-8] ZDVJGWXFXGJSIU-UHFFFAOYSA-N | $1.6\times10^{-1}$ $4.3\times10^{-1}$ $5.9\times10^{-1}$ $2.4\times10^{-1}$ $4.2\times10^{-1}$ | | Wang et al. (2017) Wang et al. (2017) Wang et al. (2017) Gharagheizi et al. (2012) Yaws et al. (1997) | Q Q Q Q Q | 80, 238 80, 239 80, 240 |
| 2-methyl-3-hexanol $C_7H_{16}O$ [617-29-8] RGRUUTLDBCWYBL-UHFFFAOYSA-N | $5.8\times10^{-1}$ | | Yaws et al. (1997) | Q | |
| 3-methyl-3-hexanol $C_7H_{16}O$ [597-96-6] KYWJZCSJMOILIZ-UHFFFAOYSA-N | $9.1\times10^{-2}$ $4.7\times10^{-1}$ $2.6\times10^{-1}$ $9.2\times10^{-1}$ $7.7\times10^{-1}$ | | Wang et al. (2017) Wang et al. (2017) Wang et al. (2017) Yaffe et al. (2003) Yaws et al. (1997) | Q Q Q Q Q | 80, 238 80, 239 80, 240 248, 272 |
| 4-methyl-3-hexanol $C_7H_{16}O$ [615-29-2] NZPGYIBESMMUFU-UHFFFAOYSA-N | $1.1\times10^{-1}$ $1.0\times10^{-1}$ $1.3\times10^{-1}$ $5.2\times10^{-1}$ | | Yaws (2003) Gharagheizi et al. (2012) Gharagheizi et al. (2010) Yaws et al. (1997) | X Q Q Q | 237 246 |





Table A3.2: Alcohols (ROH) (...continued)

| Substance Formula (Trivial Name) [CAS Registry Number] InChIKey | $H_s^{cp}$ (at $T^\ominus$) $\left[\dfrac{\mathrm{mol}}{\mathrm{m^3\,Pa}}\right]$ | $\dfrac{\mathrm{d}\ln H_s^{cp}}{\mathrm{d}(1/T)}$ [K] | Reference | Type | Note |
|---|---|---|---|---|---|
| 5-methyl-3-hexanol C$_7$H$_{16}$O [623-55-2] RGCZULIFYUPTAR-UHFFFAOYSA-N | $1.6\times10^{-1}$ $1.5\times10^{-1}$ $1.2\times10^{-1}$ $5.4\times10^{-1}$ | | Yaws (2003) Gharagheizi et al. (2012) Gharagheizi et al. (2010) Yaws et al. (1997) | X Q Q Q | 237 246 |
| 2-ethyl-1-pentanol C$_7$H$_{16}$O [27522-11-8] UKFQWAVMIMCNEH-UHFFFAOYSA-N | $1.0\times10^{-1}$ $1.1\times10^{-1}$ $1.1\times10^{-1}$ $3.4\times10^{-1}$ | | Yaws (2003) Gharagheizi et al. (2012) Gharagheizi et al. (2010) Yaws et al. (1997) | X Q Q Q | 237 246 |
| 3-ethyl-1-pentanol C$_7$H$_{16}$O [66225-51-2] DVEFUHVVWJONKR-UHFFFAOYSA-N | $9.1\times10^{-2}$ $1.7\times10^{-1}$ $1.2\times10^{-1}$ $3.4\times10^{-1}$ | | Yaws (2003) Gharagheizi et al. (2012) Gharagheizi et al. (2010) Yaws et al. (1997) | X Q Q Q | 237 246 |
| 3-ethyl-2-pentanol C$_7$H$_{16}$O [609-27-8] NEHRITNOSGFGGS-UHFFFAOYSA-N | $1.1\times10^{-1}$ $1.7\times10^{-1}$ $1.2\times10^{-1}$ $4.9\times10^{-1}$ | | Yaws (2003) Gharagheizi et al. (2012) Gharagheizi et al. (2010) Yaws et al. (1997) | X Q Q Q | 237 246 |
| 3-ethyl-3-pentanol C$_7$H$_{16}$O [597-49-9] XKIRHOWVQWCYBT-UHFFFAOYSA-N | $3.0\times10^{-1}$ 1.2 1.1 | | Plyasunov and Shock (2000) Yaffe et al. (2003) Yaws et al. (1997) | L Q Q | 248, 249 |
| 2,2-dimethyl-1-pentanol C$_7$H$_{16}$O [2370-12-9] QTOMCRXZFDHJOL-UHFFFAOYSA-N | $9.6\times10^{-2}$ $5.4\times10^{-2}$ $1.1\times10^{-1}$ $3.3\times10^{-1}$ | | Yaws (2003) Gharagheizi et al. (2012) Gharagheizi et al. (2010) Yaws et al. (1997) | X Q Q Q | 237 246 |
| 2,3-dimethyl-1-pentanol C$_7$H$_{16}$O [10143-23-4] MIBBFRQOCRYDDB-UHFFFAOYSA-N | $3.6\times10^{-1}$ | | Yaws et al. (1997) | Q | |
| 2,4-dimethyl-1-pentanol C$_7$H$_{16}$O [6305-71-1] OVOVDHYEOQJKMD-UHFFFAOYSA-N | $9.4\times10^{-2}$ $9.5\times10^{-2}$ $1.0\times10^{-1}$ $3.3\times10^{-1}$ | | Yaws (2003) Gharagheizi et al. (2012) Gharagheizi et al. (2010) Yaws et al. (1997) | X Q Q Q | 237 246 |
| 3,3-dimethyl-1-pentanol C$_7$H$_{16}$O [19264-94-9] IFDHMBHPLKHMOY-UHFFFAOYSA-N | $3.5\times10^{-1}$ | | Yaws et al. (1997) | Q | |
| 3,4-dimethyl-1-pentanol C$_7$H$_{16}$O [6570-87-2] SVJNECJVNWTYQG-UHFFFAOYSA-N | $1.6\times10^{-1}$ $1.6\times10^{-1}$ $1.1\times10^{-1}$ $3.5\times10^{-1}$ | | Yaws (2003) Gharagheizi et al. (2012) Gharagheizi et al. (2010) Yaws et al. (1997) | X Q Q Q | 237 246 |





Table A3.2: Alcohols (ROH) (...continued)

| Substance<br>Formula<br>(Trivial Name)<br>[CAS Registry Number]<br>InChIKey | $H_s^{cp}$<br>(at $T^{\ominus}$)<br>$\left[\dfrac{\text{mol}}{\text{m}^3\,\text{Pa}}\right]$ | $\dfrac{\text{d}\ln H_s^{cp}}{\text{d}(1/T)}$<br><br>[K] | Reference | Type | Note |
|---|---|---|---|---|---|
| 4,4-dimethyl-1-pentanol<br>$C_7H_{16}O$<br>[3121-79-7]<br>OWCNPTHAWPMOJU-UHFFFAOYSA-N | $8.5\times10^{-2}$<br>$1.4\times10^{-1}$<br>$1.1\times10^{-1}$<br>$3.8\times10^{-1}$ | | Yaws (2003)<br>Gharagheizi et al. (2012)<br>Gharagheizi et al. (2010)<br>Yaws et al. (1997) | X<br>Q<br>Q<br>Q | 237<br><br>246<br> |
| 2,3-dimethyl-2-pentanol<br>$C_7H_{16}O$<br>[4911-70-0]<br>YRSIFCHKXFKNME-UHFFFAOYSA-N | $9.2\times10^{-1}$<br>$8.6\times10^{-1}$ | | Yaffe et al. (2003)<br>Yaws et al. (1997) | Q<br>Q | 248, 249<br> |
| 2,4-dimethyl-2-pentanol<br>$C_7H_{16}O$<br>[625-06-9]<br>FMLSQAUAAGVTJO-UHFFFAOYSA-N | $4.0\times10^{-1}$<br>$5.7\times10^{-1}$ | | Yaffe et al. (2003)<br>Yaws et al. (1997) | Q<br>Q | 248, 272<br> |
| 3,3-dimethyl-2-pentanol<br>$C_7H_{16}O$<br>[19781-24-9]<br>FCUOIGTWJDBTHR-UHFFFAOYSA-N | $9.0\times10^{-2}$<br>$1.0\times10^{-1}$<br>$1.2\times10^{-1}$<br>$5.5\times10^{-1}$ | | Yaws (2003)<br>Gharagheizi et al. (2012)<br>Gharagheizi et al. (2010)<br>Yaws et al. (1997) | X<br>Q<br>Q<br>Q | 237<br><br>246<br> |
| 3,4-dimethyl-2-pentanol<br>$C_7H_{16}O$<br>[64502-86-9]<br>SODKTKJZYBOMMT-UHFFFAOYSA-N | $1.8\times10^{-1}$<br>$4.7\times10^{-1}$ | | Gharagheizi et al. (2012)<br>Yaws et al. (1997) | Q<br>Q | |
| 4,4-dimethyl-2-pentanol<br>$C_7H_{16}O$<br>[6144-93-0]<br>OIBKGNPMOMMSSI-UHFFFAOYSA-N | $6.8\times10^{-1}$ | | Yaws et al. (1997) | Q | |
| 2,2-dimethyl-3-pentanol<br>$C_7H_{16}O$<br>[3970-62-5]<br>HMSVXZJWPVIVIV-UHFFFAOYSA-N | $4.0\times10^{-1}$<br>$4.0\times10^{-1}$ | | Yaffe et al. (2003)<br>Yaws et al. (1997) | Q<br>Q | 248, 249<br> |
| 2,3-dimethyl-3-pentanol<br>$C_7H_{16}O$<br>[595-41-5]<br>RFZHJHSNHYIRNE-UHFFFAOYSA-N | $9.2\times10^{-1}$<br>$9.2\times10^{-1}$ | | Yaffe et al. (2003)<br>Yaws et al. (1997) | Q<br>Q | 248, 249<br> |
| 2,4-dimethyl-3-pentanol<br>$C_7H_{16}O$<br>[600-36-2]<br>BAYAKMPRFGNNFW-UHFFFAOYSA-N | $4.0\times10^{-1}$<br>$3.8\times10^{-1}$ | | Yaffe et al. (2003)<br>Yaws et al. (1997) | Q<br>Q | 248, 249<br> |
| 2-ethyl-2-methyl-1-butanol<br>$C_7H_{16}O$<br>[18371-13-6]<br>KMWHQWJVSALJAW-UHFFFAOYSA-N | $4.3\times10^{-1}$ | | Yaws et al. (1997) | Q | |



Table A3.2: Alcohols (ROH) (. . . continued)

| Substance Formula (Trivial Name) [CAS Registry Number] InChIKey | $H_s^{cp}$ (at $T^\ominus$) $\left[\dfrac{\text{mol}}{\text{m}^3\,\text{Pa}}\right]$ | $\dfrac{\text{d}\ln H_s^{cp}}{\text{d}(1/T)}$ [K] | Reference | Type | Note |
|---|---|---|---|---|---|
| 2-ethyl-3-methyl-1-butanol $C_7H_{16}O$ [32444-34-1] OXFFPTMSBXBVLZ-UHFFFAOYSA-N | $3.8\times10^{-1}$ | | Yaws et al. (1997) | Q | |
| 2,2,3-trimethyl-1-butanol $C_7H_{16}O$ [55505-23-2] FAMOAJMEFGYSNY-UHFFFAOYSA-N | $8.3\times10^{-2}$ $6.2\times10^{-2}$ $1.0\times10^{-1}$ $4.3\times10^{-1}$ | | Yaws (2003) Gharagheizi et al. (2012) Gharagheizi et al. (2010) Yaws et al. (1997) | X Q Q Q | 237 246 |
| 2,3,3-trimethyl-1-butanol $C_7H_{16}O$ [36794-64-6] IWWVOLGZHKFWKK-UHFFFAOYSA-N | $7.9\times10^{-2}$ $1.0\times10^{-1}$ $1.0\times10^{-1}$ $4.0\times10^{-1}$ | | Yaws (2003) Gharagheizi et al. (2012) Gharagheizi et al. (2010) Yaws et al. (1997) | X Q Q Q | 237 246 |
| 2,3,3-trimethyl-2-butanol $C_7H_{16}O$ [594-83-2] OKXVARYIKDXAEO-UHFFFAOYSA-N | $2.7\times10^{-1}$ | | Yaws et al. (1997) | Q | |
| 1-octanol $C_8H_{18}O$ [111-87-5] KBPLFHHGFOOTCA-UHFFFAOYSA-N | $3.4\times10^{-1}$ | 9000 | Brockbank (2013) | L | 1 |
| | $4.1\times10^{-1}$ | 8900 | Plyasunov and Shock (2000) | L | |
| | $1.7\times10^{-1}$ | 7000 | Wu et al. (2022a) | M | |
| | $2.1\times10^{-1}$ | 6900 | Shunthirasingham et al. (2013) | M | |
| | $1.9\times10^{-1}$ | 6000 | Lei et al. (2007) | M | 395 |
| | $6.5\times10^{-1}$ | | Altschuh et al. (1999) | M | |
| | $2.7\times10^{-1}$ | | Eger et al. (1999) | M | 14 |
| | $4.0\times10^{-1}$ | | Buttery et al. (1969) | M | |
| | $3.8\times10^{-1}$ | | Mackay et al. (2006c) | V | |
| | $2.4\times10^{-1}$ | | Mackay et al. (1995) | V | |
| | | 8900 | Abraham (1984) | V | |
| | $4.1\times10^{-1}$ | | Hine and Mookerjee (1975) | V | |
| | $4.1\times10^{-1}$ | | Butler et al. (1935) | V | |
| | $3.9\times10^{-1}$ | | Yaws (2003) | X | 258 |
| | $3.6\times10^{-1}$ | | Dupeux et al. (2022) | Q | 259 |
| | $3.8\times10^{-1}$ | | Keshavarz et al. (2022) | Q | |
| | 1.4 | | Duchowicz et al. (2020) | Q | 299 |
| | $3.3\times10^{-1}$ | | Savary et al. (2014) | Q | |
| | $4.1\times10^{-1}$ | | Li et al. (2014) | Q | 241 |
| | $2.5\times10^{-1}$ | | Raventos-Duran et al. (2010) | Q | 242, 243 |
| | $2.0\times10^{-1}$ | | Raventos-Duran et al. (2010) | Q | 244 |
| | $3.1\times10^{-1}$ | | Raventos-Duran et al. (2010) | Q | 245 |
| | $2.5\times10^{-1}$ | | Hilal et al. (2008) | Q | |
| | $5.6\times10^{-1}$ | | Modarresi et al. (2007) | Q | 67 |
| | | 8600 | Kühne et al. (2005) | Q | |
| | $3.2\times10^{-1}$ | | Yaffe et al. (2003) | Q | 248, 272 |
| | $2.6\times10^{-1}$ | | Yao et al. (2002) | Q | 229 |
| | $4.0\times10^{-1}$ | | English and Carroll (2001) | Q | 230, 260 |
| | 1.2 | | Katritzky et al. (1998) | Q | |





Table A3.2: Alcohols (ROH) (...continued)

| Substance Formula (Trivial Name) [CAS Registry Number] InChIKey | $H_s^{cp}$ (at $T^{\ominus}$) $\left[\dfrac{\mathrm{mol}}{\mathrm{m^3\,Pa}}\right]$ | $\dfrac{\mathrm{d}\ln H_s^{cp}}{\mathrm{d}(1/T)}$ [K] | Reference | Type | Note |
|---|---|---|---|---|---|
| | $3.9\times10^{-1}$ | | Yaws et al. (1997) | Q | |
| | $3.0\times10^{-1}$ | | Suzuki et al. (1992) | Q | 232 |
| | $3.9\times10^{-1}$ | | Nirmalakhandan and Speece (1988) | Q | |
| | $4.0\times10^{-1}$ | | Duchowicz et al. (2020) | ? | 185, 21 |
| | | 7700 | Kühne et al. (2005) | ? | |
| | $3.9\times10^{-1}$ | | Yaws (1999) | ? | 21 |
| | $6.2\times10^{-1}$ | | Yaws and Yang (1992) | ? | 21 |
| | $4.0\times10^{-1}$ | | Abraham et al. (1990) | ? | |
| 2-octanol $C_8H_{18}O$ [123-96-6] SJWFXCIHNDVPSH-UHFFFAOYSA-N | $2.7\times10^{-1}$ | 9600 | Brockbank (2013) | L | 1 |
| | $3.5\times10^{-1}$ | | Plyasunov and Shock (2000) | L | |
| | $2.7\times10^{-1}$ | | HSDB (2015) | V | |
| | $2.7\times10^{-1}$ | | Meylan and Howard (1991) | V | |
| | $9.6\times10^{-1}$ | | Yaws (2003) | X | 258 |
| | $4.2\times10^{-1}$ | | Dupeux et al. (2022) | Q | 259 |
| | $3.8\times10^{-1}$ | | Keshavarz et al. (2022) | Q | |
| | $5.6\times10^{-1}$ | | Duchowicz et al. (2020) | Q | 299 |
| | $2.5\times10^{-1}$ | | Raventos-Duran et al. (2010) | Q | 271, 243 |
| | $1.6\times10^{-1}$ | | Raventos-Duran et al. (2010) | Q | 244 |
| | $3.1\times10^{-1}$ | | Raventos-Duran et al. (2010) | Q | 245 |
| | $1.7\times10^{-1}$ | | Hilal et al. (2008) | Q | |
| | $2.0\times10^{-1}$ | | Modarresi et al. (2007) | Q | 67 |
| | $3.0\times10^{-1}$ | | Yaws et al. (1997) | Q | |
| | $3.2\times10^{-1}$ | | Meylan and Howard (1991) | Q | |
| | $8.0\times10^{-2}$ | | Duchowicz et al. (2020) | ? | 185, 21 |
| | $9.6\times10^{-1}$ | | Yaws (1999) | ? | 21 |
| 3-octanol $C_8H_{18}O$ [589-98-0] NMRPBPVERJPACX-UHFFFAOYSA-N | $3.2\times10^{-1}$ | | Plyasunov and Shock (2000) | L | |
| | $1.2\times10^{-1}$ | 8300 | Wu et al. (2022a) | M | |
| | $1.3\times10^{-1}$ | | Yaws (2003) | X | 237 |
| | $1.1\times10^{-1}$ | | Wang et al. (2017) | Q | 80, 238 |
| | $3.2\times10^{-1}$ | | Wang et al. (2017) | Q | 80, 239 |
| | $2.3\times10^{-1}$ | | Wang et al. (2017) | Q | 80, 240 |
| | $1.2\times10^{-1}$ | | Gharagheizi et al. (2012) | Q | |
| | $7.9\times10^{-2}$ | | Gharagheizi et al. (2010) | Q | 246 |
| | $3.1\times10^{-1}$ | | Yaws et al. (1997) | Q | |
| 4-octanol $C_8H_{18}O$ [589-62-8] WOFPPJOZXUTRAU-UHFFFAOYSA-N | $1.3\times10^{-1}$ | | Gharagheizi et al. (2012) | Q | |
| | $2.9\times10^{-1}$ | | Yaws et al. (1997) | Q | |
| 2-methyl-1-heptanol $C_8H_{18}O$ [60435-70-3] QZESEQBMSFFHRY-UHFFFAOYSA-N | $9.1\times10^{-2}$ | | Yaws (2003) | X | 237 |
| | $5.7\times10^{-2}$ | | Gharagheizi et al. (2012) | Q | |
| | $7.4\times10^{-2}$ | | Gharagheizi et al. (2010) | Q | 246 |
| | $3.4\times10^{-1}$ | | Yaws et al. (1997) | Q | 406 |



Table A3.2: Alcohols (ROH) (...continued)

| Substance Formula (Trivial Name) [CAS Registry Number] InChIKey | $H_s^{cp}$ (at $T^{\ominus}$) $\left[\dfrac{\text{mol}}{\text{m}^3\,\text{Pa}}\right]$ | $\dfrac{\mathrm{d}\ln H_s^{cp}}{\mathrm{d}(1/T)}$ [K] | Reference | Type | Note |
|---|---|---|---|---|---|
| 3-methyl-1-heptanol C$_8$H$_{18}$O [1070-32-2] MUPPEBVXFKNMCI-UHFFFAOYSA-N | $8.9\times10^{-2}$ $7.5\times10^{-2}$ $2.1\times10^{-1}$ | | Yaws (2003) Gharagheizi et al. (2010) Yaws et al. (1997) | X Q Q | 237 246 |
| 4-methyl-1-heptanol C$_8$H$_{18}$O [817-91-4] LLUQZGDMUIMPTC-UHFFFAOYSA-N | $9.3\times10^{-2}$ $7.5\times10^{-2}$ $2.3\times10^{-1}$ | | Yaws (2003) Gharagheizi et al. (2010) Yaws et al. (1997) | X Q Q | 237 246 |
| 5-methyl-1-heptanol C$_8$H$_{18}$O [7212-53-5] KFARNLMRENFOHE-UHFFFAOYSA-N | $1.1\times10^{-1}$ $2.1\times10^{-1}$ | | Gharagheizi et al. (2012) Yaws et al. (1997) | Q Q | |
| 6-methyl-1-heptanol C$_8$H$_{18}$O [1653-40-3] BWDBEAQIHAEVLV-UHFFFAOYSA-N | $1.1\times10^{-1}$ $1.1\times10^{-1}$ $5.6\times10^{-1}$ $2.0\times10^{-1}$ | | Duchowicz et al. (2020) HSDB (2015) Duchowicz et al. (2020) Yaws et al. (1997) | V V Q Q | 186 |
| 2-methyl-2-heptanol C$_8$H$_{18}$O [625-25-2] ACBMYYVZWKYLIP-UHFFFAOYSA-N | $5.1\times10^{-1}$ | | Yaws et al. (1997) | Q | |
| 3-methyl-2-heptanol C$_8$H$_{18}$O [31367-46-1] SZERMVMTUUAYML-UHFFFAOYSA-N | $7.8\times10^{-2}$ $9.5\times10^{-2}$ $7.9\times10^{-2}$ $3.9\times10^{-1}$ | | Yaws (2003) Gharagheizi et al. (2012) Gharagheizi et al. (2010) Yaws et al. (1997) | X Q Q Q | 237 246 |
| 4-methyl-2-heptanol C$_8$H$_{18}$O [56298-90-9] GUHWHNUGIGOSCN-UHFFFAOYSA-N | $1.6\times10^{-1}$ $3.4\times10^{-1}$ | | Gharagheizi et al. (2012) Yaws et al. (1997) | Q Q | |
| 5-methyl-2-heptanol C$_8$H$_{18}$O [54630-50-1] FYMBAYNKBWGEIK-UHFFFAOYSA-N | $1.0\times10^{-1}$ $1.7\times10^{-1}$ $7.7\times10^{-2}$ $3.3\times10^{-1}$ | | Yaws (2003) Gharagheizi et al. (2012) Gharagheizi et al. (2010) Yaws et al. (1997) | X Q Q Q | 237 246 |
| 6-methyl-2-heptanol C$_8$H$_{18}$O [4730-22-7] FCOUHTHQYOMLJT-UHFFFAOYSA-N | $1.6\times10^{-1}$ $3.3\times10^{-1}$ | | Gharagheizi et al. (2012) Yaws et al. (1997) | Q Q | |
| 2-methyl-3-heptanol C$_8$H$_{18}$O [18720-62-2] QGVFLDUEHSIZIG-UHFFFAOYSA-N | $3.8\times10^{-1}$ | | Yaws et al. (1997) | Q | |





Table A3.2: Alcohols (ROH) (...continued)

| Substance Formula (Trivial Name) [CAS Registry Number] InChIKey | $H_s^{cp}$ (at $T^{\ominus}$) $\left[\dfrac{\mathrm{mol}}{\mathrm{m}^3\,\mathrm{Pa}}\right]$ | $\dfrac{\mathrm{d}\ln H_s^{cp}}{\mathrm{d}(1/T)}$ [K] | Reference | Type | Note |
|---|---|---|---|---|---|
| 3-methyl-3-heptanol $C_8H_{18}O$ [5582-82-1] PQOSNJHBSNZITJ-UHFFFAOYSA-N | $2.9\times10^{-1}$ | | Yaws et al. (1997) | Q | |
| 4-methyl-3-heptanol $C_8H_{18}O$ [14979-39-6] BKQICAFAUMRYLZ-UHFFFAOYSA-N | $7.4\times10^{-2}$ $4.7\times10^{-2}$ $8.2\times10^{-2}$ $5.3\times10^{-1}$ | | Yaws (2003) Gharagheizi et al. (2012) Gharagheizi et al. (2010) Yaws et al. (1997) | X Q Q Q | 237 246 |
| 5-methyl-3-heptanol $C_8H_{18}O$ [18720-65-5] SECKOSOTZOBWEI-UHFFFAOYSA-N | $8.7\times10^{-2}$ $6.6\times10^{-2}$ $7.9\times10^{-2}$ $5.6\times10^{-1}$ | | Yaws (2003) Gharagheizi et al. (2012) Gharagheizi et al. (2010) Yaws et al. (1997) | X Q Q Q | 237 246 |
| 6-methyl-3-heptanol $C_8H_{18}O$ [18720-66-6] MNBIBGDICHMQFN-UHFFFAOYSA-N | $8.8\times10^{-2}$ $7.9\times10^{-2}$ $7.9\times10^{-2}$ | | Yaws (2003) Gharagheizi et al. (2012) Gharagheizi et al. (2010) | X Q Q | 237 246 |
| 2-methyl-4-heptanol $C_8H_{18}O$ [21570-35-4] QXPLZEKPCGUWEM-UHFFFAOYSA-N | $3.9\times10^{-1}$ | | Yaws et al. (1997) | Q | |
| 3-methyl-4-heptanol $C_8H_{18}O$ [1838-73-9] JMRDKKYZLXDPLN-UHFFFAOYSA-N | $8.2\times10^{-2}$ $6.2\times10^{-2}$ $8.2\times10^{-2}$ $4.1\times10^{-1}$ | | Yaws (2003) Gharagheizi et al. (2012) Gharagheizi et al. (2010) Yaws et al. (1997) | X Q Q Q | 237 246 |
| 4-methyl-4-heptanol $C_8H_{18}O$ [598-01-6] IQXKGRKRIRMQCQ-UHFFFAOYSA-N | $4.5\times10^{-1}$ | | Yaws et al. (1997) | Q | |
| 2-ethyl-1-hexanol $C_8H_{18}O$ [104-76-7] YIWUKEYIRIRTPP-UHFFFAOYSA-N | $4.6\times10^{-1}$ $1.5\times10^{-1}$ $3.7\times10^{-1}$ $3.8\times10^{-1}$ $5.6\times10^{-1}$ $7.0\times10^{-2}$ $2.5\times10^{-1}$ $3.1\times10^{-1}$ $3.1\times10^{-1}$ $3.1\times10^{-1}$ $4.0\times10^{-1}$ $1.1\times10^{-1}$ $4.3\times10^{-1}$ $4.2\times10^{-2}$ | 7200 | Plyasunov and Shock (2000) Wu et al. (2022a) Duchowicz et al. (2020) HSDB (2015) Duchowicz et al. (2020) Gharagheizi et al. (2012) Raventos-Duran et al. (2010) Raventos-Duran et al. (2010) Raventos-Duran et al. (2010) Hilal et al. (2008) Modarresi et al. (2007) Yao et al. (2002) Yaws et al. (1997) Yaws (1999) | L M V V Q Q Q Q Q Q Q Q Q ? | 186 271, 243 244 245 67 229, 267 21 |





Table A3.2: Alcohols (ROH) (...continued)

| Substance<br>Formula<br>(Trivial Name)<br>[CAS Registry Number]<br>InChIKey | $H_\mathrm{s}^{cp}$<br>(at $T^\ominus$)<br>$\left[\dfrac{\mathrm{mol}}{\mathrm{m^3\,Pa}}\right]$ | $\dfrac{\mathrm{d}\ln H_\mathrm{s}^{cp}}{\mathrm{d}(1/T)}$<br><br>[K] | Reference | Type | Note |
|---|---|---|---|---|---|
| 3-ethyl-1-hexanol<br>$C_8H_{18}O$<br>[41065-95-6]<br>LWWJDXKGQVEZKT-UHFFFAOYSA-N | $7.0\times10^{-2}$<br>$9.5\times10^{-2}$<br>$7.5\times10^{-2}$ | | Yaws (2003)<br>Gharagheizi et al. (2012)<br>Gharagheizi et al. (2010) | X<br>Q<br>Q | 237<br><br>246 |
| 4-ethyl-1-hexanol<br>$C_8H_{18}O$<br>[66576-32-7]<br>RLGDVTUGYZAYIX-UHFFFAOYSA-N | $7.0\times10^{-2}$<br>$9.5\times10^{-2}$<br>$7.5\times10^{-2}$ | | Yaws (2003)<br>Gharagheizi et al. (2012)<br>Gharagheizi et al. (2010) | X<br>Q<br>Q | 237<br><br>246 |
| 2,2-dimethyl-1-hexanol<br>$C_8H_{18}O$<br>[2370-13-0]<br>GSSDZVRLQDXOPL-UHFFFAOYSA-N | $4.9\times10^{-1}$ | | Yaws et al. (1997) | Q | |
| 2,3-dimethyl-1-hexanol<br>$C_8H_{18}O$<br>HIYTVAULOHUDEC-UHFFFAOYSA-N | $7.0\times10^{-2}$<br>$5.5\times10^{-2}$<br>$6.8\times10^{-2}$ | | Yaws (2003)<br>Gharagheizi et al. (2012)<br>Gharagheizi et al. (2010) | X<br>Q<br>Q | 237<br><br>246 |
| 2,4-dimethyl-1-hexanol<br>$C_8H_{18}O$<br>[3965-59-1]<br>GDRBQWCGBCJTLB-UHFFFAOYSA-N | $6.9\times10^{-2}$<br>$5.7\times10^{-2}$<br>$6.8\times10^{-2}$<br>$4.6\times10^{-1}$ | | Yaws (2003)<br>Gharagheizi et al. (2012)<br>Gharagheizi et al. (2010)<br>Yaws et al. (1997) | X<br>Q<br>Q<br>Q | 237<br><br>246<br> |
| 2,5-dimethyl-1-hexanol<br>$C_8H_{18}O$<br>[6886-16-4]<br>OBOHUKJIPIBYTA-UHFFFAOYSA-N | $6.5\times10^{-2}$<br>$6.4\times10^{-2}$<br>$6.8\times10^{-2}$<br>$4.1\times10^{-1}$ | | Yaws (2003)<br>Gharagheizi et al. (2012)<br>Gharagheizi et al. (2010)<br>Yaws et al. (1997) | X<br>Q<br>Q<br>Q | 237<br><br>246<br> |
| 3,3-dimethyl-1-hexanol<br>$C_8H_{18}O$<br>[10524-70-6]<br>RAKYQWCHMSXUEG-UHFFFAOYSA-N | $5.7\times10^{-2}$<br>$1.0\times10^{-1}$<br>$7.2\times10^{-2}$ | | Yaws (2003)<br>Gharagheizi et al. (2012)<br>Gharagheizi et al. (2010) | X<br>Q<br>Q | 237<br><br>246 |
| 3,4-dimethyl-1-hexanol<br>$C_8H_{18}O$<br>[66576-57-6]<br>QVFKMROLPSPCIX-UHFFFAOYSA-N | $6.3\times10^{-2}$<br>$9.9\times10^{-2}$<br>$7.0\times10^{-2}$ | | Yaws (2003)<br>Gharagheizi et al. (2012)<br>Gharagheizi et al. (2010) | X<br>Q<br>Q | 237<br><br>246 |
| 3,5-dimethyl-1-hexanol<br>$C_8H_{18}O$<br>[13501-73-0]<br>WETBJXIDTZXCBL-UHFFFAOYSA-N | $6.1\times10^{-2}$<br>$1.1\times10^{-1}$<br>$7.0\times10^{-2}$<br>$3.6\times10^{-1}$ | | Yaws (2003)<br>Gharagheizi et al. (2012)<br>Gharagheizi et al. (2010)<br>Yaws et al. (1997) | X<br>Q<br>Q<br>Q | 237<br><br>246<br> |
| 4,4-dimethyl-1-hexanol<br>$C_8H_{18}O$<br>[6481-95-4]<br>VZWIUKUDEGQHIO-UHFFFAOYSA-N | $5.6\times10^{-2}$<br>$1.0\times10^{-1}$<br>$7.2\times10^{-2}$ | | Yaws (2003)<br>Gharagheizi et al. (2012)<br>Gharagheizi et al. (2010) | X<br>Q<br>Q | 237<br><br>246 |





Table A3.2: Alcohols (ROH) (...continued)

| Substance Formula (Trivial Name) [CAS Registry Number] InChIKey | $H_s^{cp}$ (at $T^{\ominus}$) $\left[\dfrac{\mathrm{mol}}{\mathrm{m^3\,Pa}}\right]$ | $\dfrac{\mathrm{d}\ln H_s^{cp}}{\mathrm{d}(1/T)}$ [K] | Reference | Type | Note |
|---|---|---|---|---|---|
| 4,5-dimethyl-1-hexanol $C_8H_{18}O$ [60564-76-3] QABJATQYUASJEM-UHFFFAOYSA-N | $6.2{\times}10^{-2}$ $1.0{\times}10^{-1}$ $7.0{\times}10^{-2}$ | | Yaws (2003) Gharagheizi et al. (2012) Gharagheizi et al. (2010) | X Q Q | 237  246 |
| 5,5-dimethyl-1-hexanol $C_8H_{18}O$ [2768-18-5] QYFVEEMPFRRFNN-UHFFFAOYSA-N | $5.6{\times}10^{-2}$ $1.0{\times}10^{-1}$ $7.2{\times}10^{-2}$ | | Yaws (2003) Gharagheizi et al. (2012) Gharagheizi et al. (2010) | X Q Q | 237  246 |
| 3-ethyl-2-hexanol $C_8H_{18}O$ [24448-19-9] YBQZSEZVMFENOM-UHFFFAOYSA-N | $7.7{\times}10^{-2}$ $1.0{\times}10^{-1}$ $7.9{\times}10^{-2}$ $5.6{\times}10^{-1}$ | | Yaws (2003) Gharagheizi et al. (2012) Gharagheizi et al. (2010) Yaws et al. (1997) | X Q Q Q | 237  246 |
| 4-ethyl-2-hexanol $C_8H_{18}O$ ALAPJJMGVQNJIU-UHFFFAOYSA-N | $7.7{\times}10^{-2}$ $1.5{\times}10^{-1}$ $7.7{\times}10^{-2}$ | | Yaws (2003) Gharagheizi et al. (2012) Gharagheizi et al. (2010) | X Q Q | 237  246 |
| 2,3-dimethyl-2-hexanol $C_8H_{18}O$ [19550-03-9] BFKOEFFCVFMWPF-UHFFFAOYSA-N | $7.0{\times}10^{-1}$ | | Yaws et al. (1997) | Q | |
| 2,4-dimethyl-2-hexanol $C_8H_{18}O$ [42328-76-7] RTRZHEPXWQXCCI-UHFFFAOYSA-N | $8.9{\times}10^{-1}$ | | Yaws et al. (1997) | Q | |
| 2,5-dimethyl-2-hexanol $C_8H_{18}O$ [3730-60-7] JPUIYNHIEXIFMV-UHFFFAOYSA-N | $8.5{\times}10^{-1}$ | | Yaws et al. (1997) | Q | |
| 3,3-dimethyl-2-hexanol $C_8H_{18}O$ [22025-20-3] LTTHCWFPLXSQOD-UHFFFAOYSA-N | $7.5{\times}10^{-2}$ $5.5{\times}10^{-2}$ $7.8{\times}10^{-2}$ | | Yaws (2003) Gharagheizi et al. (2012) Gharagheizi et al. (2010) | X Q Q | 237  246 |
| 3,4-dimethyl-2-hexanol $C_8H_{18}O$ [19550-05-1] LBJWJHMCZHKLQU-UHFFFAOYSA-N | $6.8{\times}10^{-2}$ $1.1{\times}10^{-1}$ $7.7{\times}10^{-2}$ $5.2{\times}10^{-1}$ | | Yaws (2003) Gharagheizi et al. (2012) Gharagheizi et al. (2010) Yaws et al. (1997) | X Q Q Q | 237  246 |
| 3,5-dimethyl-2-hexanol $C_8H_{18}O$ [66576-27-0] VGGMROOHXNLMTP-UHFFFAOYSA-N | $8.1{\times}10^{-2}$ $8.1{\times}10^{-2}$ $7.7{\times}10^{-2}$ $7.0{\times}10^{-1}$ | | Yaws (2003) Gharagheizi et al. (2012) Gharagheizi et al. (2010) Yaws et al. (1997) | X Q Q Q | 237  246 |
| 4,4-dimethyl-2-hexanol $C_8H_{18}O$ CJGWLHDFPXIKQX-UHFFFAOYSA-N | $7.2{\times}10^{-2}$ $1.3{\times}10^{-1}$ $7.4{\times}10^{-2}$ | | Yaws (2003) Gharagheizi et al. (2012) Gharagheizi et al. (2010) | X Q Q | 237  246 |





Table A3.2: Alcohols (ROH) (...continued)

| Substance Formula (Trivial Name) [CAS Registry Number] InChIKey | $H_s^{cp}$ (at $T^{\ominus}$) $\left[\dfrac{\mathrm{mol}}{\mathrm{m^3\,Pa}}\right]$ | $\dfrac{\mathrm{d}\ln H_s^{cp}}{\mathrm{d}(1/T)}$ [K] | Reference | Type | Note |
|---|---|---|---|---|---|
| 4,5-dimethyl-2-hexanol $C_8H_{18}O$ KHSOOQUMPPYGAW-UHFFFAOYSA-N | $7.9{\times}10^{-2}$ $1.3{\times}10^{-1}$ $7.5{\times}10^{-2}$ | | Yaws (2003) Gharagheizi et al. (2012) Gharagheizi et al. (2010) | X Q Q | 237 246 |
| 5,5-dimethyl-2-hexanol $C_8H_{18}O$ [31841-77-7] NMSUCVXVCHZTEH-UHFFFAOYSA-N | $6.7{\times}10^{-2}$ $1.4{\times}10^{-1}$ $7.4{\times}10^{-2}$ $6.0{\times}10^{-1}$ | | Yaws (2003) Gharagheizi et al. (2012) Gharagheizi et al. (2010) Yaws et al. (1997) | X Q Q Q | 237 246 |
| 3-ethyl-3-hexanol $C_8H_{18}O$ [597-76-2] WNDLTOTUHMHNOC-UHFFFAOYSA-N | $7.2{\times}10^{-1}$ | | Yaws et al. (1997) | Q | |
| 4-ethyl-3-hexanol $C_8H_{18}O$ [19780-44-0] BOJLCKCCKQMGKD-UHFFFAOYSA-N | $1.3{\times}10^{-1}$ $8.2{\times}10^{-2}$ $6.3{\times}10^{-1}$ | | Yaws (2003) Gharagheizi et al. (2010) Yaws et al. (1997) | X Q Q | 237 246 |
| 2,2-dimethyl-3-hexanol $C_8H_{18}O$ [4209-90-9] PFHLGQKVKALLMD-UHFFFAOYSA-N | $6.2{\times}10^{-2}$ $3.4{\times}10^{-2}$ $8.1{\times}10^{-2}$ $7.8{\times}10^{-1}$ | | Yaws (2003) Gharagheizi et al. (2012) Gharagheizi et al. (2010) Yaws et al. (1997) | X Q Q Q | 237 246 |
| 2,3-dimethyl-3-hexanol $C_8H_{18}O$ [4166-46-5] CGWJMIUMPDDHQC-UHFFFAOYSA-N | $7.4{\times}10^{-1}$ | | Yaws et al. (1997) | Q | |
| 2,4-dimethyl-3-hexanol $C_8H_{18}O$ [13432-25-2] UCRQJBCLZKHOGX-UHFFFAOYSA-N | $7.0{\times}10^{-2}$ $3.7{\times}10^{-2}$ $8.1{\times}10^{-2}$ $7.0{\times}10^{-1}$ | | Yaws (2003) Gharagheizi et al. (2012) Gharagheizi et al. (2010) Yaws et al. (1997) | X Q Q Q | 237 246 |
| 2,5-dimethyl-3-hexanol $C_8H_{18}O$ [19550-07-3] SNKTZHPOKPYBPT-UHFFFAOYSA-N | $7.2{\times}10^{-1}$ | | Yaws et al. (1997) | Q | |
| 3,4-dimethyl-3-hexanol $C_8H_{18}O$ [19550-08-4] FJXOYCIYKQJAAF-UHFFFAOYSA-N | $8.6{\times}10^{-1}$ | | Yaws et al. (1997) | Q | |
| 3,5-dimethyl-3-hexanol $C_8H_{18}O$ [4209-91-0] INMGJWCKWKKMPN-UHFFFAOYSA-N | $8.9{\times}10^{-2}$ $1.4{\times}10^{-1}$ $7.4{\times}10^{-2}$ $8.6{\times}10^{-1}$ | | Yaws (2003) Gharagheizi et al. (2012) Gharagheizi et al. (2010) Yaws et al. (1997) | X Q Q Q | 237 246 |



Table A3.2: Alcohols (ROH) (...continued)

| Substance Formula (Trivial Name) [CAS Registry Number] InChIKey | $H_s^{cp}$ (at $T^{\ominus}$) $\left[\dfrac{\mathrm{mol}}{\mathrm{m^3\,Pa}}\right]$ | $\dfrac{\mathrm{d}\ln H_s^{cp}}{\mathrm{d}(1/T)}$ [K] | Reference | Type | Note |
|---|---|---|---|---|---|
| 4,4-dimethyl-3-hexanol C$_8$H$_{18}$O [19550-09-5] BARKHVPKOOSMNW-UHFFFAOYSA-N | $7.2\times10^{-1}$ | | Yaws et al. (1997) | Q | |
| 4,5-dimethyl-3-hexanol C$_8$H$_{18}$O UAVZZCGGBCLHIP-UHFFFAOYSA-N | $8.7\times10^{-2}$ $4.9\times10^{-2}$ $7.9\times10^{-2}$ | | Yaws (2003) Gharagheizi et al. (2012) Gharagheizi et al. (2010) | X Q Q | 237 246 |
| 5,5-dimethyl-3-hexanol C$_8$H$_{18}$O [66576-31-6] NPZRPUOKIPAIEL-UHFFFAOYSA-N | $8.4\times10^{-1}$ | | Yaws et al. (1997) | Q | |
| 2-propyl-1-pentanol C$_8$H$_{18}$O [58175-57-8] LASHFHLFDRTERB-UHFFFAOYSA-N | $4.1\times10^{-1}$ | | Yaws et al. (1997) | Q | |
| 2-ethyl-2-methyl-1-pentanol C$_8$H$_{18}$O [5970-63-8] VZCOFGHOKLEMPL-UHFFFAOYSA-N | $4.3\times10^{-1}$ | | Yaws et al. (1997) | Q | |
| 2-ethyl-3-methyl-1-pentanol C$_8$H$_{18}$O ALPFTHGDLMTYRM-UHFFFAOYSA-N | $6.7\times10^{-2}$ $6.0\times10^{-2}$ $6.8\times10^{-2}$ | | Yaws (2003) Gharagheizi et al. (2012) Gharagheizi et al. (2010) | X Q Q | 237 246 |
| 2-ethyl-4-methyl-1-pentanol C$_8$H$_{18}$O [106-67-2] QCHSJPKDWOFACC-UHFFFAOYSA-N | $4.4\times10^{-1}$ | | Yaws et al. (1997) | Q | |
| 3-ethyl-2-methyl-1-pentanol C$_8$H$_{18}$O VWQDMMBTEFDWNM-UHFFFAOYSA-N | $6.9\times10^{-2}$ $5.7\times10^{-2}$ $6.8\times10^{-2}$ | | Yaws (2003) Gharagheizi et al. (2012) Gharagheizi et al. (2010) | X Q Q | 237 246 |
| 3-ethyl-3-methyl-1-pentanol C$_8$H$_{18}$O [10524-71-7] XYUCPYMMHKJJSS-UHFFFAOYSA-N | $6.3\times10^{-2}$ $8.4\times10^{-2}$ $7.2\times10^{-2}$ | | Yaws (2003) Gharagheizi et al. (2012) Gharagheizi et al. (2010) | X Q Q | 237 246 |
| 3-ethyl-4-methyl-1-pentanol C$_8$H$_{18}$O [38514-13-5] RWIFVESHBHTZEM-UHFFFAOYSA-N | $6.9\times10^{-2}$ $8.3\times10^{-2}$ $7.0\times10^{-2}$ | | Yaws (2003) Gharagheizi et al. (2012) Gharagheizi et al. (2010) | X Q Q | 237 246 |
| 2,2,3-trimethyl-1-pentanol C$_8$H$_{18}$O [57409-53-7] GBONVOIRPAJSLA-UHFFFAOYSA-N | $4.7\times10^{-1}$ | | Yaws et al. (1997) | Q | |



Table A3.2: Alcohols (ROH) (...continued)

| Substance Formula (Trivial Name) [CAS Registry Number] InChIKey | $H_s^{cp}$ (at $T^{\ominus}$) $\left[\dfrac{\text{mol}}{\text{m}^3\,\text{Pa}}\right]$ | $\dfrac{\text{d}\ln H_s^{cp}}{\text{d}(1/T)}$ [K] | Reference | Type | Note |
|---|---|---|---|---|---|
| 2,2,4-trimethyl-1-pentanol $C_8H_{18}O$ [123-44-4] CWPPDTVYIJETDF-UHFFFAOYSA-N | $5.6\times10^{-1}$ | | Yaws et al. (1997) | Q | |
| 2,3,3-trimethyl-1-pentanol $C_8H_{18}O$ LFRFDOWJBYDJGD-UHFFFAOYSA-N | $5.7\times10^{-2}$ $6.0\times10^{-2}$ $6.7\times10^{-2}$ | | Yaws (2003) Gharagheizi et al. (2012) Gharagheizi et al. (2010) | X Q Q | 237 246 |
| 2,3,4-trimethyl-1-pentanol $C_8H_{18}O$ [6570-88-3] PAZDSSMTPLLLIR-UHFFFAOYSA-N | $5.5\times10^{-2}$ $7.3\times10^{-2}$ $6.3\times10^{-2}$ $3.6\times10^{-1}$ | | Yaws (2003) Gharagheizi et al. (2012) Gharagheizi et al. (2010) Yaws et al. (1997) | X Q Q Q | 237 246 |
| 2,4,4-trimethyl-1-pentanol $C_8H_{18}O$ [16325-63-6] ZNRVRWHPZZOTIE-UHFFFAOYSA-N | $5.2\times10^{-2}$ $5.2\times10^{-1}$ | | Gharagheizi et al. (2012) Yaws et al. (1997) | Q Q | |
| 3,3,4-trimethyl-1-pentanol $C_8H_{18}O$ OIPMJJVCOSACDS-UHFFFAOYSA-N | $6.2\times10^{-2}$ $7.6\times10^{-2}$ $6.9\times10^{-2}$ | | Yaws (2003) Gharagheizi et al. (2012) Gharagheizi et al. (2010) | X Q Q | 237 246 |
| 3,4,4-trimethyl-1-pentanol $C_8H_{18}O$ [16325-64-7] FQNYRSRCONKFCN-UHFFFAOYSA-N | $6.2\times10^{-2}$ $7.6\times10^{-2}$ $6.9\times10^{-2}$ | | Yaws (2003) Gharagheizi et al. (2012) Gharagheizi et al. (2010) | X Q Q | 237 246 |
| 3-ethyl-2-methyl-2-pentanol $C_8H_{18}O$ [19780-63-3] FRUMTAZIGSJVOJ-UHFFFAOYSA-N | $7.4\times10^{-1}$ | | Yaws et al. (1997) | Q | |
| 3-ethyl-4-methyl-2-pentanol $C_8H_{18}O$ [66576-23-6] PFRFNYYIOSQSDD-UHFFFAOYSA-N | $7.6\times10^{-2}$ $9.1\times10^{-2}$ $7.7\times10^{-2}$ $6.3\times10^{-1}$ | | Yaws (2003) Gharagheizi et al. (2012) Gharagheizi et al. (2010) Yaws et al. (1997) | X Q Q Q | 237 246 |
| 2,3,3-trimethyl-2-pentanol $C_8H_{18}O$ [23171-85-9] FBWWGYIEJGQWJP-UHFFFAOYSA-N | $8.0\times10^{-2}$ $1.2\times10^{-1}$ $6.8\times10^{-2}$ $7.0\times10^{-1}$ | | Yaws (2003) Gharagheizi et al. (2012) Gharagheizi et al. (2010) Yaws et al. (1997) | X Q Q Q | 237 246 |
| 2,3,4-trimethyl-2-pentanol $C_8H_{18}O$ [66576-26-9] FTPXXARYDFWPGE-UHFFFAOYSA-N | $7.2\times10^{-2}$ $1.7\times10^{-1}$ $6.9\times10^{-2}$ $7.4\times10^{-1}$ | | Yaws (2003) Gharagheizi et al. (2012) Gharagheizi et al. (2010) Yaws et al. (1997) | X Q Q Q | 237 246 |
| 2,4,4-trimethyl-2-pentanol $C_8H_{18}O$ [690-37-9] BSYJHYLAMMJNRC-UHFFFAOYSA-N | $9.9\times10^{-1}$ | | Yaws et al. (1997) | Q | |





Table A3.2: Alcohols (ROH) (...continued)

| Substance<br>Formula<br>(Trivial Name)<br>[CAS Registry Number]<br>InChIKey | $H_s^{cp}$<br>(at $T^\ominus$)<br>$\left[\dfrac{\text{mol}}{\text{m}^3\,\text{Pa}}\right]$ | $\dfrac{\mathrm{d}\ln H_s^{cp}}{\mathrm{d}(1/T)}$<br><br>[K] | Reference | Type | Note |
|---|---|---|---|---|---|
| 3,3,4-trimethyl-2-pentanol<br>$C_8H_{18}O$<br>[19411-41-7]<br>BOBYQFFCQPQLOB-UHFFFAOYSA-N | $6.3\times10^{-2}$<br>$6.5\times10^{-2}$<br>$7.5\times10^{-2}$<br>$6.1\times10^{-1}$ | | Yaws (2003)<br>Gharagheizi et al. (2012)<br>Gharagheizi et al. (2010)<br>Yaws et al. (1997) | X<br>Q<br>Q<br>Q | 237<br><br>246 |
| 3,4,4-trimethyl-2-pentanol<br>$C_8H_{18}O$<br>[10575-56-1]<br>MXDGSCZQIMQLII-UHFFFAOYSA-N | $7.4\times10^{-1}$ | | Yaws et al. (1997) | Q | |
| 3-ethyl-2-methyl-3-pentanol<br>$C_8H_{18}O$<br>[597-05-7]<br>DMHIJUVUPKCGLJ-UHFFFAOYSA-N | $7.5\times10^{-2}$<br>$8.1\times10^{-2}$<br>$7.8\times10^{-2}$<br>$7.0\times10^{-1}$ | | Yaws (2003)<br>Gharagheizi et al. (2012)<br>Gharagheizi et al. (2010)<br>Yaws et al. (1997) | X<br>Q<br>Q<br>Q | 237<br><br>246 |
| 3-ethyl-3-methyl-2-pentanol<br>$C_8H_{18}O$<br>[66576-22-5]<br>SLBLSROGXMXPPF-UHFFFAOYSA-N | $7.5\times10^{-2}$<br>$5.5\times10^{-2}$<br>$7.8\times10^{-2}$ | | Yaws (2003)<br>Gharagheizi et al. (2012)<br>Gharagheizi et al. (2010) | X<br>Q<br>Q | 237<br><br>246 |
| 2,2,3-trimethyl-3-pentanol<br>$C_8H_{18}O$<br>[7294-05-5]<br>KLIHWYNSISMOMR-UHFFFAOYSA-N | 1.2<br>1.1 | | Yaffe et al. (2003)<br>Yaws et al. (1997) | Q<br>Q | 248, 249 |
| 2,2,4-trimethyl-3-pentanol<br>$C_8H_{18}O$<br>[5162-48-1]<br>AXINNNJHLJWMTC-UHFFFAOYSA-N | $8.9\times10^{-1}$ | | Yaws et al. (1997) | Q | |
| 2,3,4-trimethyl-3-pentanol<br>$C_8H_{18}O$<br>[3054-92-0]<br>PLSMHHUFDLYURK-UHFFFAOYSA-N | $6.4\times10^{-2}$<br>$7.6\times10^{-2}$<br>$7.1\times10^{-2}$<br>$7.6\times10^{-1}$ | | Yaws (2003)<br>Gharagheizi et al. (2012)<br>Gharagheizi et al. (2010)<br>Yaws et al. (1997) | X<br>Q<br>Q<br>Q | 237<br><br>246 |
| 3-methyl-2-(1-methylethyl)-1-butanol<br>$C_8H_{18}O$<br>[18593-92-5]<br>IDGDEUZERZKYHG-UHFFFAOYSA-N | $6.6\times10^{-2}$<br><br>$5.5\times10^{-2}$<br>$6.3\times10^{-2}$<br>$4.9\times10^{-1}$ | | Yaws (2003)<br><br>Gharagheizi et al. (2012)<br>Gharagheizi et al. (2010)<br>Yaws et al. (1997) | X<br><br>Q<br>Q<br>Q | 237<br><br><br>246 |
| 2,2,3,3-tetramethyl-1-butanol<br>$C_8H_{18}O$<br>[66576-24-7]<br>OCTRJUXIYWWVAR-UHFFFAOYSA-N | $5.0\times10^{-2}$<br>$3.8\times10^{-2}$<br>$6.7\times10^{-2}$ | | Yaws (2003)<br>Gharagheizi et al. (2012)<br>Gharagheizi et al. (2010) | X<br>Q<br>Q | 237<br><br>246 |
| 2,2-diethyl-1-butanol<br>$C_8H_{18}O$<br>[13023-60-4]<br>CBHXDVOSUKFRBE-UHFFFAOYSA-N | $6.5\times10^{-2}$<br>$3.7\times10^{-2}$<br>$7.0\times10^{-2}$ | | Yaws (2003)<br>Gharagheizi et al. (2012)<br>Gharagheizi et al. (2010) | X<br>Q<br>Q | 237<br><br>246 |



Table A3.2: Alcohols (ROH) (...continued)

| Substance<br>Formula<br>(Trivial Name)<br>[CAS Registry Number]<br>InChIKey | $H_s^{cp}$<br>(at $T^{\ominus}$)<br>$\left[\dfrac{\text{mol}}{\text{m}^3\,\text{Pa}}\right]$ | $\dfrac{\text{d}\ln H_s^{cp}}{\text{d}(1/T)}$<br><br>[K] | Reference | Type | Note |
|---|---|---|---|---|---|
| 2,3-dimethyl-2-ethyl-1-butanol | $6.0\times10^{-2}$ | | Yaws (2003) | X | 237 |
| $C_8H_{18}O$ | $3.7\times10^{-2}$ | | Gharagheizi et al. (2012) | Q | |
| YPIJDMPDVZUWJO-UHFFFAOYSA-N | $6.6\times10^{-2}$ | | Gharagheizi et al. (2010) | Q | 246 |
| 3,3-dimethyl-2-ethyl-1-butanol | $6.0\times10^{-2}$ | | Yaws (2003) | X | 237 |
| $C_8H_{18}O$ | $5.5\times10^{-2}$ | | Gharagheizi et al. (2012) | Q | |
| [66576-56-5] | $6.7\times10^{-2}$ | | Gharagheizi et al. (2010) | Q | 246 |
| WOVMYJCSCISMAU-UHFFFAOYSA-N | | | | | |
| 2-(hexyloxy)-ethanol | 3.2 | | Duchowicz et al. (2020) | V | 186 |
| $C_8H_{18}O_2$ | 8.4 | | Duchowicz et al. (2020) | Q | |
| [112-25-4] | | | | | |
| UPGSWASWQBLSKZ-UHFFFAOYSA-N | | | | | |
| 1-nonanol | $2.7\times10^{-1}$ | 9300 | Brockbank (2013) | L | 1 |
| $C_9H_{20}O$ | $1.1\times10^{-1}$ | 6300 | Shunthirasingham et al. (2013) | M | |
| [143-08-8] | $1.4\times10^{-1}$ | 6200 | Lei et al. (2007) | M | 407, 395 |
| ZWRUINPWMLAQRD-UHFFFAOYSA-N | $3.2\times10^{-1}$ | | Duchowicz et al. (2020) | V | 186 |
| | $3.2\times10^{-1}$ | | HSDB (2015) | V | |
| | $2.8\times10^{-1}$ | | Abraham (1984) | V | |
| | $3.8\times10^{-1}$ | | Yaws (2003) | X | 258 |
| | $3.5\times10^{-1}$ | | Dupeux et al. (2022) | Q | 259 |
| | $5.4\times10^{-1}$ | | Abney (2021) | Q | 399 |
| | 1.4 | | Duchowicz et al. (2020) | Q | |
| | $2.0\times10^{-1}$ | | Raventos-Duran et al. (2010) | Q | 271, 243 |
| | $1.6\times10^{-1}$ | | Raventos-Duran et al. (2010) | Q | 244 |
| | $2.5\times10^{-1}$ | | Raventos-Duran et al. (2010) | Q | 245 |
| | $2.2\times10^{-1}$ | | Hilal et al. (2008) | Q | |
| | $5.1\times10^{-1}$ | | Modarresi et al. (2007) | Q | 67 |
| | $3.2\times10^{-1}$ | | Yaffe et al. (2003) | Q | 248, 249 |
| | $2.4\times10^{-1}$ | | Yao et al. (2002) | Q | 229 |
| | 1.0 | | Katritzky et al. (1998) | Q | |
| | $3.1\times10^{-1}$ | | Nirmalakhandan et al. (1997) | Q | |
| | $3.2\times10^{-1}$ | | Yaws et al. (1997) | Q | |
| | $3.0\times10^{-1}$ | | Yaws (1999) | ? | 21 |
| | $5.9\times10^{-1}$ | | Yaws and Yang (1992) | ? | 21 |
| | $2.9\times10^{-1}$ | | Abraham et al. (1990) | ? | |
| 2-nonanol | $2.1\times10^{-1}$ | 11000 | Brockbank (2013) | L | 1 |
| $C_9H_{20}O$ | $6.5\times10^{-2}$ | | van Ruth et al. (2002) | M | 14 |
| [628-99-9] | $4.9\times10^{-1}$ | | van Ruth and Villeneuve (2002) | M | 14, 361 |
| NGDNVOAEIVQRFH-UHFFFAOYSA-N | $6.9\times10^{-2}$ | | van Ruth et al. (2001) | M | 14 |
| | $2.0\times10^{-1}$ | | Duchowicz et al. (2020) | V | 186 |
| | $5.6\times10^{-1}$ | | Duchowicz et al. (2020) | Q | |
| | $2.0\times10^{-1}$ | | Raventos-Duran et al. (2010) | Q | 242, 243 |
| | $1.2\times10^{-1}$ | | Raventos-Duran et al. (2010) | Q | 244 |
| | $2.5\times10^{-1}$ | | Raventos-Duran et al. (2010) | Q | 245 |
| | $2.2\times10^{-1}$ | | Modarresi et al. (2007) | Q | 67 |
| | $2.1\times10^{-1}$ | | Yao et al. (2002) | Q | 229 |



Table A3.2: Alcohols (ROH) (...continued)

| Substance Formula (Trivial Name) [CAS Registry Number] InChIKey | $H_s^{cp}$ (at $T^{\ominus}$) $\left[\dfrac{\text{mol}}{\text{m}^3\,\text{Pa}}\right]$ | $\dfrac{\text{d}\ln H_s^{cp}}{\text{d}(1/T)}$ [K] | Reference | Type | Note |
|---|---|---|---|---|---|
| | $5.4\times10^{-1}$ | | Yaws et al. (1997) | Q | |
| | $2.0\times10^{-1}$ | | Yaws (1999) | ? | 21, 80 |
| 3-nonanol $C_9H_{20}O$ [624-51-1] GYSCXPVAKHVAAY-UHFFFAOYSA-N | $5.2\times10^{-2}$ $9.6\times10^{-2}$ $2.6\times10^{-1}$ $2.0\times10^{-1}$ $8.4\times10^{-2}$ $5.2\times10^{-2}$ $3.0\times10^{-1}$ | | Yaws (2003) Wang et al. (2017) Wang et al. (2017) Wang et al. (2017) Gharagheizi et al. (2012) Gharagheizi et al. (2010) Yaws et al. (1997) | X Q Q Q Q Q Q | 237 80, 238 80, 239 80, 240 246 |
| 4-nonanol $C_9H_{20}O$ [5932-79-6] IXUOEGRSQCCEHB-UHFFFAOYSA-N | $5.4\times10^{-2}$ $8.0\times10^{-2}$ $5.2\times10^{-2}$ $3.1\times10^{-1}$ | | Yaws (2003) Gharagheizi et al. (2012) Gharagheizi et al. (2010) Yaws et al. (1997) | X Q Q Q | 237 246 |
| 5-nonanol $C_9H_{20}O$ [623-93-8] FCBBRODPXVPZAH-UHFFFAOYSA-N | $5.2\times10^{-2}$ $8.5\times10^{-2}$ $5.2\times10^{-2}$ $2.9\times10^{-1}$ | | Yaws (2003) Gharagheizi et al. (2012) Gharagheizi et al. (2010) Yaws et al. (1997) | X Q Q Q | 237 246 |
| 2-methyl-1-octanol $C_9H_{20}O$ [818-81-5] IGVGCQGTEINVOH-UHFFFAOYSA-N | $4.8\times10^{-2}$ $4.5\times10^{-2}$ $5.0\times10^{-2}$ | | Yaws (2003) Gharagheizi et al. (2012) Gharagheizi et al. (2010) | X Q Q | 237 246 |
| 3-methyl-1-octanol $C_9H_{20}O$ [38514-02-2] CLFSZAMBOZSCOS-UHFFFAOYSA-N | $4.8\times10^{-2}$ $6.6\times10^{-2}$ $5.0\times10^{-2}$ | | Yaws (2003) Gharagheizi et al. (2012) Gharagheizi et al. (2010) | X Q Q | 237 246 |
| 4-methyl-1-octanol $C_9H_{20}O$ [38514-03-3] MWWKESKJRHQWEF-UHFFFAOYSA-N | $4.8\times10^{-2}$ $6.6\times10^{-2}$ $5.0\times10^{-2}$ | | Yaws (2003) Gharagheizi et al. (2012) Gharagheizi et al. (2010) | X Q Q | 237 246 |
| 5-methyl-1-octanol $C_9H_{20}O$ [38514-04-4] CGCDFYUPZXVGIX-UHFFFAOYSA-N | $4.8\times10^{-2}$ $6.6\times10^{-2}$ $5.0\times10^{-2}$ | | Yaws (2003) Gharagheizi et al. (2012) Gharagheizi et al. (2010) | X Q Q | 237 246 |
| 6-methyl-1-octanol $C_9H_{20}O$ [38514-05-5] WWRGKAMABZHMCN-UHFFFAOYSA-N | $4.3\times10^{-2}$ $5.0\times10^{-2}$ $2.1\times10^{-1}$ | | Yaws (2003) Gharagheizi et al. (2010) Yaws et al. (1997) | X Q Q | 237 246 |
| 7-methyl-1-octanol $C_9H_{20}O$ [2430-22-0] QDTDKYHPHANITQ-UHFFFAOYSA-N | $4.3\times10^{-2}$ $5.0\times10^{-2}$ $2.1\times10^{-1}$ | | Yaws (2003) Gharagheizi et al. (2010) Yaws et al. (1997) | X Q Q | 237 246 |



Table A3.2: Alcohols (ROH) (...continued)

| Substance Formula (Trivial Name) [CAS Registry Number] InChIKey | $H_s^{cp}$ (at $T^\ominus$) $\left[\dfrac{\text{mol}}{\text{m}^3\,\text{Pa}}\right]$ | $\dfrac{\text{d}\ln H_s^{cp}}{\text{d}(1/T)}$ [K] | Reference | Type | Note |
|---|---|---|---|---|---|
| 2-methyl-2-octanol | $5.7\times10^{-2}$ | | Yaws (2003) | X | 237 |
| $C_9H_{20}O$ | $1.7\times10^{-1}$ | | Gharagheizi et al. (2012) | Q | |
| [628-44-4] | $5.3\times10^{-2}$ | | Gharagheizi et al. (2010) | Q | 246 |
| KBCNUEXDHWDIFX-UHFFFAOYSA-N | $4.8\times10^{-1}$ | | Yaws et al. (1997) | Q | |
| 3-methyl-2-octanol | $6.0\times10^{-2}$ | | Yaws (2003) | X | 237 |
| $C_9H_{20}O$ | $5.7\times10^{-2}$ | | Gharagheizi et al. (2012) | Q | |
| [27644-49-1] | $5.2\times10^{-2}$ | | Gharagheizi et al. (2010) | Q | 246 |
| WQADSKJNOTZWML-UHFFFAOYSA-N | | | | | |
| 5-methyl-2-octanol | $6.0\times10^{-2}$ | | Yaws (2003) | X | 237 |
| $C_9H_{20}O$ | $8.3\times10^{-2}$ | | Gharagheizi et al. (2012) | Q | |
| [66793-81-5] | $5.0\times10^{-2}$ | | Gharagheizi et al. (2010) | Q | 246 |
| IWFICVXFFNDWOJ-UHFFFAOYSA-N | | | | | |
| 7-methyl-2-octanol | $6.0\times10^{-2}$ | | Yaws (2003) | X | 237 |
| $C_9H_{20}O$ | $8.3\times10^{-2}$ | | Gharagheizi et al. (2012) | Q | |
| [66793-83-7] | $5.0\times10^{-2}$ | | Gharagheizi et al. (2010) | Q | 246 |
| NOEKZKTXHKNMAQ-UHFFFAOYSA-N | | | | | |
| 2-methyl-3-octanol | $5.7\times10^{-2}$ | | Yaws (2003) | X | 237 |
| $C_9H_{20}O$ | $4.2\times10^{-2}$ | | Gharagheizi et al. (2012) | Q | |
| [26533-34-6] | $5.5\times10^{-2}$ | | Gharagheizi et al. (2010) | Q | 246 |
| DIVBBSLQUDHECU-UHFFFAOYSA-N | $4.1\times10^{-1}$ | | Yaws et al. (1997) | Q | |
| 3-methyl-3-octanol | $5.4\times10^{-2}$ | | Yaws (2003) | X | 237 |
| $C_9H_{20}O$ | $4.8\times10^{-2}$ | | Yaws (2003) | X | 237 |
| [5340-36-3] | $1.3\times10^{-1}$ | | Gharagheizi et al. (2012) | Q | |
| JEWXYDDSLPIBBO-UHFFFAOYSA-N | $1.6\times10^{-1}$ | | Gharagheizi et al. (2012) | Q | |
| | $5.6\times10^{-2}$ | | Gharagheizi et al. (2010) | Q | 246 |
| | $5.6\times10^{-2}$ | | Gharagheizi et al. (2010) | Q | 246 |
| | $3.6\times10^{-1}$ | | Yaws et al. (1997) | Q | |
| 4-methyl-3-octanol | $5.6\times10^{-2}$ | | Yaws (2003) | X | 237 |
| $C_9H_{20}O$ | $4.5\times10^{-2}$ | | Gharagheizi et al. (2012) | Q | |
| [66793-80-4] | $5.5\times10^{-2}$ | | Gharagheizi et al. (2010) | Q | 246 |
| XPWBBDBJJRZCPW-UHFFFAOYSA-N | | | | | |
| 6-methyl-3-octanol | $5.6\times10^{-2}$ | | Yaws (2003) | X | 237 |
| $C_9H_{20}O$ | $6.6\times10^{-2}$ | | Gharagheizi et al. (2012) | Q | |
| [40225-75-0] | $5.2\times10^{-2}$ | | Gharagheizi et al. (2010) | Q | 246 |
| MFYHIHFYDULUQP-UHFFFAOYSA-N | | | | | |
| 7-methyl-3-octanol | $5.6\times10^{-2}$ | | Yaws (2003) | X | 237 |
| $C_9H_{20}O$ | $6.6\times10^{-2}$ | | Gharagheizi et al. (2012) | Q | |
| [66793-84-8] | $5.2\times10^{-2}$ | | Gharagheizi et al. (2010) | Q | 246 |
| IDCFJIMYNKBKMB-UHFFFAOYSA-N | | | | | |



Table A3.2: Alcohols (ROH) (...continued)

| Substance Formula (Trivial Name) [CAS Registry Number] InChIKey | $H_s^{cp}$ (at $T^\ominus$) $\left[\dfrac{\mathrm{mol}}{\mathrm{m}^3\,\mathrm{Pa}}\right]$ | $\dfrac{\mathrm{d}\ln H_s^{cp}}{\mathrm{d}(1/T)}$ [K] | Reference | Type | Note |
|---|---|---|---|---|---|
| 2-methyl-4-octanol | $5.7\times10^{-2}$ | | Yaws (2003) | X | 237 |
| $C_9H_{20}O$ | $6.2\times10^{-2}$ | | Gharagheizi et al. (2012) | Q | |
| [40575-41-5] | $5.2\times10^{-2}$ | | Gharagheizi et al. (2010) | Q | 246 |
| BIAVIOIDPRPYJK-UHFFFAOYSA-N | $4.1\times10^{-1}$ | | Yaws et al. (1997) | Q | |
| 3-methyl-4-octanol | $6.1\times10^{-2}$ | | Yaws (2003) | X | 237 |
| $C_9H_{20}O$ | $3.8\times10^{-2}$ | | Gharagheizi et al. (2012) | Q | |
| [26533-35-7] | $5.5\times10^{-2}$ | | Gharagheizi et al. (2010) | Q | 246 |
| MJOKZMZDONULOD-UHFFFAOYSA-N | $4.6\times10^{-1}$ | | Yaws et al. (1997) | Q | |
| 4-methyl-4-octanol | $5.5\times10^{-2}$ | | Yaws (2003) | X | 237 |
| $C_9H_{20}O$ | $1.2\times10^{-1}$ | | Gharagheizi et al. (2012) | Q | |
| [23418-37-3] | $5.6\times10^{-2}$ | | Gharagheizi et al. (2010) | Q | 246 |
| RXSIKQJQLQRQQY-UHFFFAOYSA-N | $4.5\times10^{-1}$ | | Yaws et al. (1997) | Q | |
| 5-methyl-4-octanol | $5.9\times10^{-2}$ | | Yaws (2003) | X | 237 |
| $C_9H_{20}O$ | $4.0\times10^{-2}$ | | Gharagheizi et al. (2012) | Q | |
| [59734-23-5] | $5.5\times10^{-2}$ | | Gharagheizi et al. (2010) | Q | 246 |
| YLTHHPQUTLMNIF-UHFFFAOYSA-N | | | | | |
| 6-methyl-4-octanol | $5.9\times10^{-2}$ | | Yaws (2003) | X | 237 |
| $C_9H_{20}O$ | $5.8\times10^{-2}$ | | Gharagheizi et al. (2012) | Q | |
| [66793-82-6] | $5.2\times10^{-2}$ | | Gharagheizi et al. (2010) | Q | 246 |
| KFRCBGHGDZSOJV-UHFFFAOYSA-N | | | | | |
| 7-methyl-4-octanol | $5.9\times10^{-2}$ | | Yaws (2003) | X | 237 |
| $C_9H_{20}O$ | $5.8\times10^{-2}$ | | Gharagheizi et al. (2012) | Q | |
| [33933-77-6] | $5.2\times10^{-2}$ | | Gharagheizi et al. (2010) | Q | 246 |
| KJMBBHZOLRRVMV-UHFFFAOYSA-N | | | | | |
| 2-ethyl-1-heptanol | $4.9\times10^{-2}$ | | Yaws (2003) | X | 237 |
| $C_9H_{20}O$ | $4.4\times10^{-2}$ | | Gharagheizi et al. (2012) | Q | |
| [817-60-7] | $5.0\times10^{-2}$ | | Gharagheizi et al. (2010) | Q | 246 |
| QNJAZNNWHWYOEO-UHFFFAOYSA-N | | | | | |
| 3-ethyl-1-heptanol | $4.2\times10^{-2}$ | | Yaws (2003) | X | 237 |
| $C_9H_{20}O$ | $5.0\times10^{-2}$ | | Gharagheizi et al. (2010) | Q | 246 |
| [3525-25-5] | $2.0\times10^{-1}$ | | Yaws et al. (1997) | Q | |
| VRZRVMXNGMZLDB-UHFFFAOYSA-N | | | | | |
| 5-ethyl-1-heptanol | $4.9\times10^{-2}$ | | Yaws (2003) | X | 237 |
| $C_9H_{20}O$ | $5.0\times10^{-2}$ | | Gharagheizi et al. (2010) | Q | 246 |
| [998-65-2] | | | | | |
| GOJBFVJUMYSMJJ-UHFFFAOYSA-N | | | | | |
| 3-ethyl-2-heptanol | $5.9\times10^{-2}$ | | Yaws (2003) | X | 237 |
| $C_9H_{20}O$ | $7.1\times10^{-2}$ | | Gharagheizi et al. (2012) | Q | |
| [19780-39-3] | $5.8\times10^{-2}$ | | Gharagheizi et al. (2012) | Q | |
| MMQDVLWWGWJSFS-UHFFFAOYSA-N | $5.2\times10^{-2}$ | | Gharagheizi et al. (2010) | Q | 246 |



Table A3.2: Alcohols (ROH) (...continued)

| Substance Formula (Trivial Name) [CAS Registry Number] InChIKey | $H_s^{cp}$ (at $T^{\ominus}$) $\left[\dfrac{\mathrm{mol}}{\mathrm{m^3\,Pa}}\right]$ | $\dfrac{\mathrm{d}\ln H_s^{cp}}{\mathrm{d}(1/T)}$ [K] | Reference | Type | Note |
|---|---|---|---|---|---|
| 3-ethyl-3-heptanol<br>$C_9H_{20}O$<br>[19780-41-7]<br>XKRZDNKKANUBPV-UHFFFAOYSA-N | $5.4\times10^{-2}$<br>$8.7\times10^{-2}$<br>$5.9\times10^{-2}$<br>$4.3\times10^{-1}$ | | Yaws (2003)<br>Gharagheizi et al. (2012)<br>Gharagheizi et al. (2010)<br>Yaws et al. (1997) | X<br>Q<br>Q<br>Q | 237<br><br>246 |
| 4-ethyl-4-heptanol<br>$C_9H_{20}O$<br>[597-90-0]<br>GNROHGFUVTWFNG-UHFFFAOYSA-N | $5.7\times10^{-2}$<br>$7.9\times10^{-2}$<br>$5.9\times10^{-2}$<br>$4.7\times10^{-1}$ | | Yaws (2003)<br>Gharagheizi et al. (2012)<br>Gharagheizi et al. (2010)<br>Yaws et al. (1997) | X<br>Q<br>Q<br>Q | 237<br><br>246 |
| 2,2-dimethyl-1-heptanol<br>$C_9H_{20}O$<br>[14250-79-4]<br>WENIXZFPXMQPQQ-UHFFFAOYSA-N | $4.5\times10^{-2}$<br>$2.5\times10^{-2}$<br>$4.8\times10^{-2}$<br>$3.2\times10^{-1}$ | | Yaws (2003)<br>Gharagheizi et al. (2012)<br>Gharagheizi et al. (2010)<br>Yaws et al. (1997) | X<br>Q<br>Q<br>Q | 237<br><br>246 |
| 4,6-dimethyl-1-heptanol<br>$C_9H_{20}O$<br>[820-05-3]<br>GCBXGQPBCBPHSP-UHFFFAOYSA-N | $5.9\times10^{-2}$<br>$4.0\times10^{-2}$<br>$4.6\times10^{-2}$ | | Yaws (2003)<br>Gharagheizi et al. (2012)<br>Gharagheizi et al. (2010) | X<br>Q<br>Q | 237<br><br>246 |
| 6,6-dimethyl-1-heptanol<br>$C_9H_{20}O$<br>[65769-10-0]<br>IOFUAVGBFVXDAO-UHFFFAOYSA-N | $5.4\times10^{-2}$<br>$4.0\times10^{-2}$<br>$4.7\times10^{-2}$ | | Yaws (2003)<br>Gharagheizi et al. (2012)<br>Gharagheizi et al. (2010) | X<br>Q<br>Q | 237<br><br>246 |
| 2,3-dimethyl-2-heptanol<br>$C_9H_{20}O$<br>[66794-00-1]<br>PQSMEVPHTJECDZ-UHFFFAOYSA-N | $5.0\times10^{-2}$<br>$1.3\times10^{-1}$<br>$5.2\times10^{-2}$ | | Yaws (2003)<br>Gharagheizi et al. (2012)<br>Gharagheizi et al. (2010) | X<br>Q<br>Q | 237<br><br>246 |
| 2,4-dimethyl-2-heptanol<br>$C_9H_{20}O$<br>[65822-93-7]<br>VORBOMMQBCSRQF-UHFFFAOYSA-N | $5.0\times10^{-2}$<br>$5.0\times10^{-2}$ | | Yaws (2003)<br>Gharagheizi et al. (2010) | X<br>Q | 237<br>246 |
| 2,5-dimethyl-2-heptanol<br>$C_9H_{20}O$<br>[1561-18-8]<br>LVIFBEPHIBJBEU-UHFFFAOYSA-N | $5.0\times10^{-2}$<br>$5.0\times10^{-2}$ | | Yaws (2003)<br>Gharagheizi et al. (2010) | X<br>Q | 237<br>246 |
| 2,6-dimethyl-2-heptanol<br>$C_9H_{20}O$<br>[13254-34-7]<br>HGDVHRITTGWMJK-UHFFFAOYSA-N | $5.9\times10^{-2}$<br>$1.4\times10^{-1}$<br>$5.0\times10^{-2}$<br>$5.8\times10^{-1}$ | | Yaws (2003)<br>Gharagheizi et al. (2012)<br>Gharagheizi et al. (2010)<br>Yaws et al. (1997) | X<br>Q<br>Q<br>Q | 237<br><br>246 |
| 4,6-dimethyl-2-heptanol<br>$C_9H_{20}O$<br>[51079-52-8]<br>YYUGBYFBCFRGNZ-UHFFFAOYSA-N | $4.4\times10^{-2}$<br>$1.2\times10^{-1}$<br>$4.9\times10^{-2}$<br>$3.1\times10^{-1}$ | | Yaws (2003)<br>Gharagheizi et al. (2012)<br>Gharagheizi et al. (2010)<br>Yaws et al. (1997) | X<br>Q<br>Q<br>Q | 237<br><br>246 |



Table A3.2: Alcohols (ROH) (... continued)

| Substance Formula (Trivial Name) [CAS Registry Number] InChIKey | $H_s^{cp}$ (at $T^{\ominus}$) $\left[\dfrac{\text{mol}}{\text{m}^3\,\text{Pa}}\right]$ | $\dfrac{\text{d}\ln H_s^{cp}}{\text{d}(1/T)}$ [K] | Reference | Type | Note |
|---|---|---|---|---|---|
| 5,6-dimethyl-2-heptanol $C_9H_{20}O$ [58795-24-7] ONBXNGYZXZBRJA-UHFFFAOYSA-N | $4.6\times10^{-2}$ $1.2\times10^{-1}$ $4.9\times10^{-2}$ $3.2\times10^{-1}$ | | Yaws (2003) Gharagheizi et al. (2012) Gharagheizi et al. (2010) Yaws et al. (1997) | X Q Q Q | 237 246 |
| 2,2-dimethyl-3-heptanol $C_9H_{20}O$ [19549-70-3] QENWAAAGDINHGE-UHFFFAOYSA-N | $5.3\times10^{-2}$ $2.5\times10^{-2}$ $5.5\times10^{-2}$ | | Yaws (2003) Gharagheizi et al. (2012) Gharagheizi et al. (2010) | X Q Q | 237 246 |
| 2,3-dimethyl-3-heptanol $C_9H_{20}O$ [19549-71-4] JIEGVNXCNNWVPH-UHFFFAOYSA-N | $5.7\times10^{-2}$ $7.0\times10^{-2}$ $5.4\times10^{-2}$ $5.4\times10^{-1}$ | | Yaws (2003) Gharagheizi et al. (2012) Gharagheizi et al. (2010) Yaws et al. (1997) | X Q Q Q | 237 246 |
| 2,6-dimethyl-3-heptanol $C_9H_{20}O$ [19549-73-6] XZDMJRIWJSNEGC-UHFFFAOYSA-N | $6.1\times10^{-2}$ $3.3\times10^{-2}$ $5.4\times10^{-2}$ $5.2\times10^{-1}$ | | Yaws (2003) Gharagheizi et al. (2012) Gharagheizi et al. (2010) Yaws et al. (1997) | X Q Q Q | 237 246 |
| 3,5-dimethyl-3-heptanol $C_9H_{20}O$ [19549-74-7] NOSOEGQGQOMHBF-UHFFFAOYSA-N | $5.4\times10^{-2}$ $1.1\times10^{-1}$ $5.2\times10^{-2}$ | | Yaws (2003) Gharagheizi et al. (2012) Gharagheizi et al. (2010) | X Q Q | 237 246 |
| 2,2-dimethyl-4-heptanol $C_9H_{20}O$ [66793-99-5] ZIKNXBYJTDTRQA-UHFFFAOYSA-N | $5.7\times10^{-2}$ $4.7\times10^{-2}$ $5.0\times10^{-2}$ $5.4\times10^{-1}$ | | Yaws (2003) Gharagheizi et al. (2012) Gharagheizi et al. (2010) Yaws et al. (1997) | X Q Q Q | 237 246 |
| 2,4-dimethyl-4-heptanol $C_9H_{20}O$ [19549-77-0] QKRRAXACNUNGCF-UHFFFAOYSA-N | $5.9\times10^{-2}$ $9.5\times10^{-2}$ $5.2\times10^{-2}$ $5.7\times10^{-1}$ | | Yaws (2003) Gharagheizi et al. (2012) Gharagheizi et al. (2010) Yaws et al. (1997) | X Q Q Q | 237 246 |
| 2,5-dimethyl-4-heptanol $C_9H_{20}O$ NXLQTSHQSAVKJB-UHFFFAOYSA-N | $1.6\times10^{-1}$ | | Yaws et al. (1997) | Q | |
| 2,6-dimethyl-4-heptanol $C_9H_{20}O$ [108-82-7] HXQPUEQDBSPXTE-UHFFFAOYSA-N | $1.0\times10^{-1}$ $7.7\times10^{-2}$ $8.6\times10^{-2}$ $2.0\times10^{-1}$ $1.6\times10^{-1}$ $2.5\times10^{-1}$ $1.7\times10^{-1}$ $3.7\times10^{-1}$ $1.6\times10^{-1}$ $1.7\times10^{-1}$ $1.7\times10^{-1}$ | 9300 | Brockbank (2013) Duchowicz et al. (2020) Duchowicz et al. (2020) Raventos-Duran et al. (2010) Raventos-Duran et al. (2010) Raventos-Duran et al. (2010) Hilal et al. (2008) Modarresi et al. (2007) Katritzky et al. (1998) Yaws et al. (1997) Yaws (1999) | L V Q Q Q Q Q Q Q Q ? | 1 186 242, 243 244 245 67 21 |



Table A3.2: Alcohols (ROH) (...continued)

| Substance Formula (Trivial Name) [CAS Registry Number] InChIKey | $H_s^{cp}$ (at $T^{\ominus}$) $\left[\dfrac{\text{mol}}{\text{m}^3\,\text{Pa}}\right]$ | $\dfrac{\mathrm{d}\ln H_s^{cp}}{\mathrm{d}(1/T)}$ [K] | Reference | Type | Note |
|---|---|---|---|---|---|
| 3,3-dimethyl-4-heptanol $C_9H_{20}O$ [19549-78-1] UGAZGYFCTFGQQH-UHFFFAOYSA-N | $5.5\times10^{-2}$ $2.3\times10^{-2}$ $5.5\times10^{-2}$ $2.2\times10^{-1}$ | | Yaws (2003) Gharagheizi et al. (2012) Gharagheizi et al. (2010) Yaws et al. (1997) | X Q Q Q | 237 246 |
| 3,5-dimethyl-4-heptanol $C_9H_{20}O$ [19549-79-2] ZKXITRNXHWEQJU-UHFFFAOYSA-N | $5.0\times10^{-2}$ $3.2\times10^{-2}$ $5.6\times10^{-2}$ | | Yaws (2003) Gharagheizi et al. (2012) Gharagheizi et al. (2010) | X Q Q | 237 246 |
| 3,6-dimethyl-3-heptanol $C_9H_{20}O$ [1573-28-0] HVPGGLNDHUWMLS-UHFFFAOYSA-N | $6.8\times10^{-2}$ $1.0\times10^{-1}$ $5.2\times10^{-2}$ | | Yaws (2003) Gharagheizi et al. (2012) Gharagheizi et al. (2010) | X Q Q | 237 246 |
| 2-ethyl-3-methyl-1-hexanol $C_9H_{20}O$ [66794-04-5] DXXCMAAACWSRPE-UHFFFAOYSA-N | $4.9\times10^{-2}$ $3.7\times10^{-2}$ $4.6\times10^{-2}$ | | Yaws (2003) Gharagheizi et al. (2012) Gharagheizi et al. (2010) | X Q Q | 237 246 |
| 2-ethyl-4-methyl-1-hexanol $C_9H_{20}O$ [66794-06-7] NRZVENBFUFCASY-UHFFFAOYSA-N | $4.7\times10^{-2}$ $4.0\times10^{-2}$ $4.6\times10^{-2}$ $1.2\times10^{-1}$ | | Yaws (2003) Gharagheizi et al. (2012) Gharagheizi et al. (2010) Yaws et al. (1997) | X Q Q Q | 237 246 |
| 2-ethyl-5-methyl-1-hexanol $C_9H_{20}O$ [66794-07-8] KVJOUCNJVQBRAO-UHFFFAOYSA-N | $4.9\times10^{-2}$ $3.7\times10^{-2}$ $4.6\times10^{-2}$ | | Yaws (2003) Gharagheizi et al. (2012) Gharagheizi et al. (2010) | X Q Q | 237 246 |
| 3-ethyl-1-methyl-1-hexanol $C_9H_{20}O$ YNZQSDTXBBBFQH-UHFFFAOYSA-N | $5.0\times10^{-2}$ $1.2\times10^{-1}$ $5.0\times10^{-2}$ | | Yaws (2003) Gharagheizi et al. (2012) Gharagheizi et al. (2010) | X Q Q | 237 246 |
| 3-ethyl-2-methyl-1-hexanol $C_9H_{20}O$ [66794-01-2] UDZCUJOKBNABRS-UHFFFAOYSA-N | $1.3\times10^{-1}$ | | Yaws et al. (1997) | Q | |
| 3,3,5-trimethyl-1-hexanol $C_9H_{20}O$ [1484-87-3] HRXNWQMMTLLJJQ-UHFFFAOYSA-N | $4.1\times10^{-2}$ $5.6\times10^{-2}$ $4.5\times10^{-2}$ | | Yaws (2003) Gharagheizi et al. (2012) Gharagheizi et al. (2010) | X Q Q | 237 246 |
| 3,4,4-trimethyl-1-hexanol $C_9H_{20}O$ [66793-73-5] OLRPKAJSAOYYCY-UHFFFAOYSA-N | $4.2\times10^{-2}$ $5.3\times10^{-2}$ $4.5\times10^{-2}$ $1.4\times10^{-1}$ | | Yaws (2003) Gharagheizi et al. (2012) Gharagheizi et al. (2010) Yaws et al. (1997) | X Q Q Q | 237 246 |



Table A3.2: Alcohols (ROH) (...continued)

| Substance Formula (Trivial Name) [CAS Registry Number] InChIKey | $H_s^{cp}$ (at $T^\ominus$) $\left[\dfrac{\text{mol}}{\text{m}^3\,\text{Pa}}\right]$ | $\dfrac{\text{d}\ln H_s^{cp}}{\text{d}(1/T)}$ [K] | Reference | Type | Note |
|---|---|---|---|---|---|
| 3,5,5-trimethyl-1-hexanol C$_9$H$_{20}$O [3452-97-9] BODRLKRKPXBDBN-UHFFFAOYSA-N | $4.1\times10^{-2}$ $5.6\times10^{-2}$ $4.5\times10^{-2}$ $1.3\times10^{-1}$ | | Yaws (2003) Gharagheizi et al. (2012) Gharagheizi et al. (2010) Yaws et al. (1997) | X Q Q Q | 237 246 |
| 4,5,5-trimethyl-1-hexanol C$_9$H$_{20}$O [66793-75-7] IECJUMYEWJOKHM-UHFFFAOYSA-N | $3.4\times10^{-2}$ $7.3\times10^{-2}$ $4.5\times10^{-2}$ $9.4\times10^{-2}$ | | Yaws (2003) Gharagheizi et al. (2012) Gharagheizi et al. (2010) Yaws et al. (1997) | X Q Q Q | 237 246 |
| 2-propyl-1-hexanol C$_9$H$_{20}$O [817-46-9] JSUXZEJWGVYJJG-UHFFFAOYSA-N | $5.8\times10^{-2}$ $3.3\times10^{-2}$ $5.0\times10^{-2}$ | | Yaws (2003) Gharagheizi et al. (2012) Gharagheizi et al. (2010) | X Q Q | 237 246 |
| 3-propyl-1-hexanol C$_9$H$_{20}$O [66793-85-9] MHJIVMDBOGBUHH-UHFFFAOYSA-N | $5.8\times10^{-2}$ $4.8\times10^{-2}$ $5.0\times10^{-2}$ | | Yaws (2003) Gharagheizi et al. (2012) Gharagheizi et al. (2010) | X Q Q | 237 246 |
| 3-ethyl-2-methyl-2-hexanol C$_9$H$_{20}$O [66794-02-3] HTOYXTGJEOUKPM-UHFFFAOYSA-N | $5.3\times10^{-2}$ $1.2\times10^{-1}$ $5.2\times10^{-2}$ $2.1\times10^{-1}$ | | Yaws (2003) Gharagheizi et al. (2012) Gharagheizi et al. (2010) Yaws et al. (1997) | X Q Q Q | 237 246 |
| 2,3,4-trimethyl-2-hexanol C$_9$H$_{20}$O [21102-13-6] JTOONBKGYWVPOY-UHFFFAOYSA-N | $3.7\times10^{-2}$ $4.8\times10^{-2}$ | | Yaws (2003) Gharagheizi et al. (2010) | X Q | 237 246 |
| 2,4,4-trimethyl-2-hexanol C$_9$H$_{20}$O [66793-91-7] QMEHJTXRYVSFES-UHFFFAOYSA-N | $3.4\times10^{-2}$ $4.5\times10^{-2}$ | | Yaws (2003) Gharagheizi et al. (2010) | X Q | 237 246 |
| 2,4,5-trimethyl-2-hexanol C$_9$H$_{20}$O [66793-93-9] CIVZBWYMPDUPSQ-UHFFFAOYSA-N | $3.7\times10^{-2}$ $4.7\times10^{-2}$ | | Yaws (2003) Gharagheizi et al. (2010) | X Q | 237 246 |
| 2,5,5-trimethyl-2-hexanol C$_9$H$_{20}$O [66793-71-3] ASOCSFZHADJKFU-UHFFFAOYSA-N | $3.4\times10^{-2}$ $4.5\times10^{-2}$ | | Yaws (2003) Gharagheizi et al. (2010) | X Q | 237 246 |
| 3-ethyl-2-methyl-3-hexanol C$_9$H$_{20}$O [66794-03-4] VUMFTWNZYFBYEB-UHFFFAOYSA-N | $4.8\times10^{-2}$ $6.4\times10^{-2}$ $5.5\times10^{-2}$ $1.7\times10^{-1}$ | | Yaws (2003) Gharagheizi et al. (2012) Gharagheizi et al. (2010) Yaws et al. (1997) | X Q Q Q | 237 246 |



Table A3.2: Alcohols (ROH) (...continued)

| Substance Formula (Trivial Name) [CAS Registry Number] InChIKey | $H_s^{cp}$ (at $T^{\ominus}$) $\left[\dfrac{\mathrm{mol}}{\mathrm{m}^3\,\mathrm{Pa}}\right]$ | $\dfrac{\mathrm{d}\ln H_s^{cp}}{\mathrm{d}(1/T)}$ [K] | Reference | Type | Note |
|---|---|---|---|---|---|
| 3-ethyl-4-methyl-3-hexanol C$_9$H$_{20}$O [51200-80-7] YOCLWWFBEJQTBM-UHFFFAOYSA-N | $5.2\times10^{-2}$ $5.5\times10^{-2}$ $5.5\times10^{-2}$ | | Yaws (2003) Gharagheizi et al. (2012) Gharagheizi et al. (2010) | X Q Q | 237 246 |
| 3-ethyl-5-methyl-3-hexanol C$_9$H$_{20}$O [597-77-3] RMBUNHOAKTUXTC-UHFFFAOYSA-N | $5.8\times10^{-2}$ $6.6\times10^{-2}$ $5.4\times10^{-2}$ $2.5\times10^{-1}$ | | Yaws (2003) Gharagheizi et al. (2012) Gharagheizi et al. (2010) Yaws et al. (1997) | X Q Q Q | 237 246 |
| 4-ethyl-2-methyl-3-hexanol C$_9$H$_{20}$O [33943-21-4] GBUYIUMHXYBONU-UHFFFAOYSA-N | $5.7\times10^{-2}$ $2.5\times10^{-2}$ $5.6\times10^{-2}$ | | Yaws (2003) Gharagheizi et al. (2012) Gharagheizi et al. (2010) | X Q Q | 237 246 |
| 4-ethyl-3-methyl-3-hexanol C$_9$H$_{20}$O [66794-05-6] DUGZCTGTSJAPCV-UHFFFAOYSA-N | $5.2\times10^{-2}$ $8.0\times10^{-2}$ $5.4\times10^{-2}$ | | Yaws (2003) Gharagheizi et al. (2012) Gharagheizi et al. (2010) | X Q Q | 237 246 |
| 2,2,3-trimethyl-3-hexanol C$_9$H$_{20}$O [5340-41-0] DHNHHYWBZNQHGN-UHFFFAOYSA-N | $4.8\times10^{-2}$ $4.7\times10^{-2}$ $5.0\times10^{-2}$ $2.4\times10^{-1}$ | | Yaws (2003) Gharagheizi et al. (2012) Gharagheizi et al. (2010) Yaws et al. (1997) | X Q Q Q | 237 246 |
| 2,2,4-trimethyl-3-hexanol C$_9$H$_{20}$O [66793-89-3] YMBVVPXERVAJTB-UHFFFAOYSA-N | $5.7\times10^{-2}$ $5.5\times10^{-2}$ $2.7\times10^{-1}$ | | Yaws (2003) Gharagheizi et al. (2010) Yaws et al. (1997) | X Q Q | 237 246 |
| 2,2,5-trimethyl-3-hexanol C$_9$H$_{20}$O [3970-60-3] UCLXUUYVWJKXIK-UHFFFAOYSA-N | $6.6\times10^{-2}$ $5.3\times10^{-2}$ $3.4\times10^{-1}$ | | Yaws (2003) Gharagheizi et al. (2010) Yaws et al. (1997) | X Q Q | 237 246 |
| 2,3,4-trimethyl-3-hexanol C$_9$H$_{20}$O [66793-90-6] LTKHKGAPVGSECA-UHFFFAOYSA-N | $6.0\times10^{-2}$ $5.0\times10^{-2}$ | | Yaws (2003) Gharagheizi et al. (2010) | X Q | 237 246 |
| 2,3,5-trimethyl-3-hexanol C$_9$H$_{20}$O [65927-60-8] ZVKQNIUBLJYSGN-UHFFFAOYSA-N | $6.0\times10^{-2}$ $4.9\times10^{-2}$ | | Yaws (2003) Gharagheizi et al. (2010) | X Q | 237 246 |
| 2,4,4-trimethyl-3-hexanol C$_9$H$_{20}$O [66793-92-8] PDIVDRVTPZNRDZ-UHFFFAOYSA-N | $5.5\times10^{-2}$ $5.5\times10^{-2}$ $2.5\times10^{-1}$ | | Yaws (2003) Gharagheizi et al. (2010) Yaws et al. (1997) | X Q Q | 237 246 |





Table A3.2: Alcohols (ROH) (...continued)

| Substance<br>Formula<br>(Trivial Name)<br>[CAS Registry Number]<br>InChIKey | $H_s^{cp}$<br>(at $T^\ominus$)<br>$\left[\dfrac{\mathrm{mol}}{\mathrm{m^3\,Pa}}\right]$ | $\dfrac{\mathrm{d}\ln H_s^{cp}}{\mathrm{d}(1/T)}$<br><br>[K] | Reference | Type | Note |
|---|---|---|---|---|---|
| 2,5,5-trimethyl-3-hexanol<br>$C_9H_{20}O$<br>[66793-72-4]<br>NPUIBFBNZSKEAP-UHFFFAOYSA-N | $5.7\times10^{-2}$<br>$2.8\times10^{-2}$<br>$5.1\times10^{-2}$ | | Yaws (2003)<br>Gharagheizi et al. (2012)<br>Gharagheizi et al. (2010) | X<br>Q<br>Q | 237<br><br>246 |
| 3,4,4-trimethyl-3-hexanol<br>$C_9H_{20}O$<br>[66793-74-6]<br>QWJDXMFJOMPTCV-UHFFFAOYSA-N | $5.5\times10^{-2}$<br>$5.0\times10^{-2}$<br>$2.9\times10^{-1}$ | | Yaws (2003)<br>Gharagheizi et al. (2010)<br>Yaws et al. (1997) | X<br>Q<br>Q | 237<br>246 |
| 3,5,5-trimethyl-3-hexanol<br>$C_9H_{20}O$<br>[66810-87-5]<br>HGGJNDQMXCTKCR-UHFFFAOYSA-N | $5.5\times10^{-2}$<br>$4.7\times10^{-2}$ | | Yaws (2003)<br>Gharagheizi et al. (2010) | X<br>Q | 237<br>246 |
| 4-methyl-2-propyl-1-pentanol<br>$C_9H_{20}O$<br>[54004-41-0]<br>IGSWOIOCVJEQRH-UHFFFAOYSA-N | $5.0\times10^{-2}$<br>$3.8\times10^{-2}$<br>$4.6\times10^{-2}$<br>$1.3\times10^{-1}$ | | Yaws (2003)<br>Gharagheizi et al. (2012)<br>Gharagheizi et al. (2010)<br>Yaws et al. (1997) | X<br>Q<br>Q<br>Q | 237<br><br>246 |
| 4-methyl-2-(1-methylethyl)-1-pentanol<br>$C_9H_{20}O$<br>[55505-24-3]<br>CBHPZSADBWMIGE-UHFFFAOYSA-N | $5.0\times10^{-2}$<br><br>$3.1\times10^{-2}$<br>$4.3\times10^{-2}$<br>$1.6\times10^{-1}$ | | Yaws (2003)<br><br>Gharagheizi et al. (2012)<br>Gharagheizi et al. (2010)<br>Yaws et al. (1997) | X<br><br>Q<br>Q<br>Q | 237<br><br><br>246 |
| 2-ethyl-2,4-dimethyl-1-pentanol<br>$C_9H_{20}O$<br>[66793-98-4]<br>JIZZVMCXCCJUHI-UHFFFAOYSA-N | $4.4\times10^{-2}$<br>$2.2\times10^{-2}$<br>$4.5\times10^{-2}$<br>$1.5\times10^{-1}$ | | Yaws (2003)<br>Gharagheizi et al. (2012)<br>Gharagheizi et al. (2010)<br>Yaws et al. (1997) | X<br>Q<br>Q<br>Q | 237<br><br>246 |
| 3-ethyl-2,2-dimethyl-1-pentanol<br>$C_9H_{20}O$<br>[66793-95-1]<br>HEQYMZTUSVDQBW-UHFFFAOYSA-N | $4.5\times10^{-2}$<br>$2.2\times10^{-2}$<br>$4.5\times10^{-2}$ | | Yaws (2003)<br>Gharagheizi et al. (2012)<br>Gharagheizi et al. (2010) | X<br>Q<br>Q | 237<br><br>246 |
| 2,3,3,4-tetramethyl-2-pentanol<br>$C_9H_{20}O$<br>[66793-86-0]<br>HLWCPAJEXXTJHY-UHFFFAOYSA-N | $3.9\times10^{-2}$<br>$8.6\times10^{-2}$<br>$4.4\times10^{-2}$ | | Yaws (2003)<br>Gharagheizi et al. (2012)<br>Gharagheizi et al. (2010) | X<br>Q<br>Q | 237<br><br>246 |
| 2,3,4,4-tetramethyl-2-pentanol<br>$C_9H_{20}O$<br>[66793-87-1]<br>DBCCRNKDJIOKIO-UHFFFAOYSA-N | $3.9\times10^{-2}$<br>$1.3\times10^{-1}$<br>$4.3\times10^{-2}$ | | Yaws (2003)<br>Gharagheizi et al. (2012)<br>Gharagheizi et al. (2010) | X<br>Q<br>Q | 237<br><br>246 |
| 3,3,4,4-tetramethyl-2-pentanol<br>$C_9H_{20}O$<br>[66793-88-2]<br>AIQOKYGLKGGWOI-UHFFFAOYSA-N | $3.9\times10^{-2}$<br>$4.0\times10^{-2}$<br>$4.8\times10^{-2}$<br>$2.0\times10^{-1}$ | | Yaws (2003)<br>Gharagheizi et al. (2012)<br>Gharagheizi et al. (2010)<br>Yaws et al. (1997) | X<br>Q<br>Q<br>Q | 237<br><br>246 |





Table A3.2: Alcohols (ROH) (...continued)

| Substance<br>Formula<br>(Trivial Name)<br>[CAS Registry Number]<br>InChIKey | $H_s^{cp}$<br>(at $T^{\ominus}$)<br>$\left[\dfrac{\text{mol}}{\text{m}^3\,\text{Pa}}\right]$ | $\dfrac{\text{d}\ln H_s^{cp}}{\text{d}(1/T)}$<br><br>[K] | Reference | Type | Note |
|---|---|---|---|---|---|
| 3,3-diethyl-2-pentanol<br>$C_9H_{20}O$<br>[66793-94-0]<br>IENXYHLNFHBNRR-UHFFFAOYSA-N | $4.5\times10^{-2}$<br>$4.8\times10^{-2}$<br>$5.3\times10^{-2}$ | | Yaws (2003)<br>Gharagheizi et al. (2012)<br>Gharagheizi et al. (2010) | X<br>Q<br>Q | 237<br><br>246 |
| 2,3-dimethyl-3-ethyl-2-pentanol<br>$C_9H_{20}O$<br>[66793-97-3]<br>AVQHNUMJCWRCBA-UHFFFAOYSA-N | $3.7\times10^{-2}$<br>$1.1\times10^{-1}$<br>$4.8\times10^{-2}$ | | Yaws (2003)<br>Gharagheizi et al. (2012)<br>Gharagheizi et al. (2010) | X<br>Q<br>Q | 237<br><br>246 |
| 4,4-dimethyl-3-ethyl-2-pentanol<br>$C_9H_{20}O$<br>[21102-09-0]<br>YGQDLQQRBCOMKJ-UHFFFAOYSA-N | $4.1\times10^{-2}$<br>$7.2\times10^{-2}$<br>$4.9\times10^{-2}$ | | Yaws (2003)<br>Gharagheizi et al. (2012)<br>Gharagheizi et al. (2010) | X<br>Q<br>Q | 237<br><br>246 |
| 3-ethyl-2,2-dimethyl-3-pentanol<br>$C_9H_{20}O$<br>[66793-96-2]<br>CFWIFHZJKFFDFU-UHFFFAOYSA-N | $4.8\times10^{-2}$<br>$3.3\times10^{-2}$<br>$5.1\times10^{-2}$<br>$2.3\times10^{-1}$ | | Yaws (2003)<br>Gharagheizi et al. (2012)<br>Gharagheizi et al. (2010)<br>Yaws et al. (1997) | X<br>Q<br>Q<br>Q | 237<br><br>246<br> |
| 3-ethyl-2,4-dimethyl-3-pentanol<br>$C_9H_{20}O$<br>[3970-59-0]<br>PZPYIRQFCOFLGJ-UHFFFAOYSA-N | $4.9\times10^{-2}$<br>$4.8\times10^{-2}$<br>$3.7\times10^{-2}$<br>$3.6\times10^{-2}$<br>$5.1\times10^{-2}$<br>$5.1\times10^{-2}$<br>$2.1\times10^{-1}$ | | Yaws (2003)<br>Yaws (2003)<br>Gharagheizi et al. (2012)<br>Gharagheizi et al. (2012)<br>Gharagheizi et al. (2010)<br>Gharagheizi et al. (2010)<br>Yaws et al. (1997) | X<br>X<br>Q<br>Q<br>Q<br>Q<br>Q | 237<br>237<br><br><br>246<br>246<br> |
| 2,2,3,4-tetramethyl-3-pentanol<br>$C_9H_{20}O$<br>[29772-39-2]<br>KNTFAYXQHRNXSM-UHFFFAOYSA-N | $4.4\times10^{-2}$<br>$3.4\times10^{-2}$<br>$4.6\times10^{-2}$<br>$2.3\times10^{-1}$ | | Yaws (2003)<br>Gharagheizi et al. (2012)<br>Gharagheizi et al. (2010)<br>Yaws et al. (1997) | X<br>Q<br>Q<br>Q | 237<br><br>246<br> |
| 2,2,4,4-tetramethyl-3-pentanol<br>$C_9H_{20}O$<br>[14609-79-1]<br>WFJSIIHYYLHRHB-UHFFFAOYSA-N | $5.0\times10^{-2}$<br>$5.4\times10^{-2}$<br>$2.8\times10^{-1}$ | | Yaws (2003)<br>Gharagheizi et al. (2010)<br>Yaws et al. (1997) | X<br>Q<br>Q | 237<br>246<br> |
| 1-decanol<br>$C_{10}H_{22}O$<br>[112-30-1]<br>MWKFXSUHUHTGQN-UHFFFAOYSA-N | $2.0\times10^{-1}$<br>$7.6\times10^{-2}$<br>$6.5\times10^{-2}$<br>$3.1\times10^{-1}$<br>$1.9\times10^{-1}$<br>$2.1\times10^{-1}$<br>$2.8\times10^{-1}$<br>$1.3\times10^{-1}$<br>$1.5$<br>$1.6\times10^{-1}$<br>$1.2\times10^{-1}$<br>$2.0\times10^{-1}$<br>$2.0\times10^{-1}$ | 9200<br>6600<br>5300 | Brockbank (2013)<br>Shunthirasingham et al. (2013)<br>Lei et al. (2007)<br>Altschuh et al. (1999)<br>Abraham (1984)<br>Yaws (2003)<br>Dupeux et al. (2022)<br>Keshavarz et al. (2022)<br>Duchowicz et al. (2020)<br>Raventos-Duran et al. (2010)<br>Raventos-Duran et al. (2010)<br>Raventos-Duran et al. (2010)<br>Hilal et al. (2008) | L<br>M<br>M<br>M<br>V<br>X<br>Q<br>Q<br>Q<br>Q<br>Q<br>Q<br>Q | 1, 408<br><br>395<br><br><br>258<br>259<br><br><br>271, 243<br>244<br>245<br> |



Table A3.2: Alcohols (ROH) (...continued)

| Substance Formula (Trivial Name) [CAS Registry Number] InChIKey | $H_s^{cp}$ (at $T^\ominus$) $\left[\dfrac{\text{mol}}{\text{m}^3\,\text{Pa}}\right]$ | $\dfrac{\text{d}\ln H_s^{cp}}{\text{d}(1/T)}$ [K] | Reference | Type | Note |
|---|---|---|---|---|---|
| | $4.6\times10^{-1}$ | | Modarresi et al. (2007) | Q | 67 |
| | $2.1\times10^{-1}$ | | Yaffe et al. (2003) | Q | 248, 249 |
| | $2.2\times10^{-1}$ | | Yao et al. (2002) | Q | 229 |
| | $9.9\times10^{-1}$ | | Katritzky et al. (1998) | Q | |
| | $2.4\times10^{-1}$ | | Nirmalakhandan et al. (1997) | Q | |
| | $2.1\times10^{-1}$ | | Yaws et al. (1997) | Q | |
| | $3.1\times10^{-1}$ | | Duchowicz et al. (2020) | ? | 185, 21 |
| | $4.4\times10^{-2}$ | | Maniere et al. (2011) | ? | 241, 165 |
| | $2.1\times10^{-1}$ | | Yaws (1999) | ? | 21 |
| | $3.7\times10^{-1}$ | | Yaws and Yang (1992) | ? | 21 |
| | $1.9\times10^{-1}$ | | Abraham et al. (1990) | ? | |
| 2-decanol C$_{10}$H$_{22}$O [1120-06-5] ACUZDYFTRHEKOS-UHFFFAOYSA-N | $5.4\times10^{-1}$ | | Yaws et al. (1997) | Q | |
| 3-decanol C$_{10}$H$_{22}$O [1565-81-7] ICEQLCZWZXUUIJ-UHFFFAOYSA-N | $3.8\times10^{-2}$ $7.8\times10^{-2}$ $2.1\times10^{-1}$ $2.0\times10^{-1}$ $7.9\times10^{-2}$ $3.5\times10^{-2}$ | | Yaws (2003) Wang et al. (2017) Wang et al. (2017) Wang et al. (2017) Gharagheizi et al. (2012) Gharagheizi et al. (2010) | X Q Q Q Q Q | 237 80, 238 80, 239 80, 240 246 |
| 4-decanol C$_{10}$H$_{22}$O [2051-31-2] DTDMYWXTWWFLGJ-UHFFFAOYSA-N | $3.8\times10^{-2}$ $7.9\times10^{-2}$ $3.5\times10^{-2}$ $5.3\times10^{-1}$ | | Yaws (2003) Gharagheizi et al. (2012) Gharagheizi et al. (2010) Yaws et al. (1997) | X Q Q Q | 237 246 |
| 5-decanol C$_{10}$H$_{22}$O [5205-34-5] SZMNDOUFZGODBR-UHFFFAOYSA-N | $4.5\times10^{-2}$ $6.1\times10^{-2}$ $3.5\times10^{-2}$ $7.3\times10^{-1}$ | | Yaws (2003) Gharagheizi et al. (2012) Gharagheizi et al. (2010) Yaws et al. (1997) | X Q Q Q | 237 246 |
| 2-methyl-1-nonanol C$_{10}$H$_{22}$O [40589-14-8] BEGNRPGEHZBNKK-UHFFFAOYSA-N | $3.0\times10^{-2}$ $3.6\times10^{-2}$ $3.4\times10^{-1}$ | | Yaws (2003) Gharagheizi et al. (2010) Yaws et al. (1997) | X Q Q | 237 246 |
| 2-methyl-2-nonanol C$_{10}$H$_{22}$O [10297-57-1] VREDNSVJXRJXRI-UHFFFAOYSA-N | $2.9\times10^{-2}$ $3.9\times10^{-2}$ | | Yaws (2003) Gharagheizi et al. (2010) | X Q | 237 246 |
| 2-methyl-3-nonanol C$_{10}$H$_{22}$O [26533-33-5] OFIYMUXECPHIPZ-UHFFFAOYSA-N | $3.6\times10^{-2}$ $3.4\times10^{-2}$ $3.9\times10^{-2}$ $5.7\times10^{-1}$ | | Yaws (2003) Gharagheizi et al. (2012) Gharagheizi et al. (2010) Yaws et al. (1997) | X Q Q Q | 237 246 |



Table A3.2: Alcohols (ROH) (...continued)

| Substance Formula (Trivial Name) [CAS Registry Number] InChIKey | $H_s^{cp}$ (at $T^{\ominus}$) $\left[\dfrac{\text{mol}}{\text{m}^3\,\text{Pa}}\right]$ | $\dfrac{\text{d}\ln H_s^{cp}}{\text{d}(1/T)}$ [K] | Reference | Type | Note |
|---|---|---|---|---|---|
| 2-methyl-4-nonanol $C_{10}H_{22}O$ [26533-31-3] IBHHTADZYDLHPM-UHFFFAOYSA-N | $3.6\times10^{-2}$ $5.1\times10^{-2}$ $3.6\times10^{-2}$ | | Yaws (2003) Gharagheizi et al. (2012) Gharagheizi et al. (2010) | X Q Q | 237 246 |
| 2-methyl-5-nonanol $C_{10}H_{22}O$ [29843-62-7] RLQVUGAVOCBRNQ-UHFFFAOYSA-N | $3.6\times10^{-2}$ $5.1\times10^{-2}$ $3.6\times10^{-2}$ | | Yaws (2003) Gharagheizi et al. (2012) Gharagheizi et al. (2010) | X Q Q | 237 246 |
| 3-methyl-1-nonanol $C_{10}H_{22}O$ [22663-64-5] BXQPYGLPOMTAPU-UHFFFAOYSA-N | $3.5\times10^{-2}$ $4.0\times10^{-2}$ $3.5\times10^{-2}$ | | Yaws (2003) Gharagheizi et al. (2012) Gharagheizi et al. (2010) | X Q Q | 237 246 |
| 3-methyl-2-nonanol $C_{10}H_{22}O$ [60671-32-1] QCBDLZKDRUKOFS-UHFFFAOYSA-N | $3.2\times10^{-2}$ $5.9\times10^{-2}$ $3.6\times10^{-2}$ | | Yaws (2003) Gharagheizi et al. (2012) Gharagheizi et al. (2010) | X Q Q | 237 246 |
| 3-methyl-3-nonanol $C_{10}H_{22}O$ [21078-72-8] VZBFPIMCUSPDLS-UHFFFAOYSA-N | $3.2\times10^{-2}$ $1.1\times10^{-1}$ $4.1\times10^{-2}$ | | Yaws (2003) Gharagheizi et al. (2012) Gharagheizi et al. (2010) | X Q Q | 237 246 |
| 3-methyl-5-nonanol $C_{10}H_{22}O$ QRZAYHZIHWIJHD-UHFFFAOYSA-N | $3.6\times10^{-2}$ $5.1\times10^{-2}$ $3.6\times10^{-2}$ | | Yaws (2003) Gharagheizi et al. (2012) Gharagheizi et al. (2010) | X Q Q | 237 246 |
| 4-methyl-1-nonanol $C_{10}H_{22}O$ [1489-47-0] HECVGHIJHFNAIL-UHFFFAOYSA-N | $3.1\times10^{-2}$ $4.2\times10^{-2}$ $3.5\times10^{-2}$ | | Yaws (2003) Gharagheizi et al. (2012) Gharagheizi et al. (2010) | X Q Q | 237 246 |
| 4-methyl-4-nonanol $C_{10}H_{22}O$ [23418-38-4] GDCOAKPWVJCNGI-UHFFFAOYSA-N | $3.2\times10^{-2}$ $1.1\times10^{-1}$ $4.1\times10^{-2}$ | | Yaws (2003) Gharagheizi et al. (2012) Gharagheizi et al. (2010) | X Q Q | 237 246 |
| 5-methyl-1-nonanol $C_{10}H_{22}O$ [2768-16-3] DBJSFYCKLLBKGB-UHFFFAOYSA-N | $3.5\times10^{-2}$ $4.0\times10^{-2}$ $3.5\times10^{-2}$ | | Yaws (2003) Gharagheizi et al. (2012) Gharagheizi et al. (2010) | X Q Q | 237 246 |
| 5-methyl-2-nonanol $C_{10}H_{22}O$ [66731-95-1] KZTLXVUOEODDMR-UHFFFAOYSA-N | $3.2\times10^{-2}$ $8.7\times10^{-2}$ $3.4\times10^{-2}$ | | Yaws (2003) Gharagheizi et al. (2012) Gharagheizi et al. (2010) | X Q Q | 237 246 |





Table A3.2: Alcohols (ROH) (...continued)

| Substance<br>Formula<br>(Trivial Name)<br>[CAS Registry Number]<br>InChIKey | $H_s^{cp}$<br>(at $T^\ominus$)<br><br>$\left[\dfrac{\text{mol}}{\text{m}^3\,\text{Pa}}\right]$ | $\dfrac{\text{d}\ln H_s^{cp}}{\text{d}(1/T)}$<br><br>[K] | Reference | Type | Note |
|---|---|---|---|---|---|
| 5-methyl-3-nonanol<br>$C_{10}H_{22}O$<br>[66719-43-5]<br>CCSZLOPBDJWYHV-UHFFFAOYSA-N | $3.6\times10^{-2}$<br>$5.1\times10^{-2}$<br>$3.6\times10^{-2}$ | | Yaws (2003)<br>Gharagheizi et al. (2012)<br>Gharagheizi et al. (2010) | X<br>Q<br>Q | 237<br><br>246 |
| 5-methyl-4-nonanol<br>$C_{10}H_{22}O$<br>[66719-44-6]<br>BFYIAYZULSQBFQ-UHFFFAOYSA-N | $3.6\times10^{-2}$<br>$3.5\times10^{-2}$<br>$3.9\times10^{-2}$ | | Yaws (2003)<br>Gharagheizi et al. (2012)<br>Gharagheizi et al. (2010) | X<br>Q<br>Q | 237<br><br>246 |
| 5-methyl-5-nonanol<br>$C_{10}H_{22}O$<br>[33933-78-7]<br>AGSIGVZAVLOKLP-UHFFFAOYSA-N | $4.9\times10^{-2}$<br>$9.1\times10^{-2}$<br>$4.1\times10^{-2}$ | | Yaws (2003)<br>Gharagheizi et al. (2012)<br>Gharagheizi et al. (2010) | X<br>Q<br>Q | 237<br><br>246 |
| 6-methyl-2-nonanol<br>$C_{10}H_{22}O$<br>[66256-60-8]<br>OYTRLQQQIHEJFE-UHFFFAOYSA-N | $3.2\times10^{-2}$<br>$8.7\times10^{-2}$<br>$3.4\times10^{-2}$ | | Yaws (2003)<br>Gharagheizi et al. (2012)<br>Gharagheizi et al. (2010) | X<br>Q<br>Q | 237<br><br>246 |
| 7-methyl-1-nonanol<br>$C_{10}H_{22}O$<br>[33234-93-4]<br>BJKXZCGJAOWNTN-UHFFFAOYSA-N | $3.5\times10^{-2}$<br>$4.0\times10^{-2}$<br>$3.5\times10^{-2}$ | | Yaws (2003)<br>Gharagheizi et al. (2012)<br>Gharagheizi et al. (2010) | X<br>Q<br>Q | 237<br><br>246 |
| 7-methyl-2-nonanol<br>$C_{10}H_{22}O$<br>[66256-61-9]<br>GCOBJHFUTQUABB-UHFFFAOYSA-N | $3.2\times10^{-2}$<br>$3.4\times10^{-2}$ | | Yaws (2003)<br>Gharagheizi et al. (2010) | X<br>Q | 237<br>246 |
| 7-methyl-4-nonanol<br>$C_{10}H_{22}O$<br>[26981-98-6]<br>KLZAWUYSXUGWGQ-UHFFFAOYSA-N | $3.6\times10^{-2}$<br>$5.1\times10^{-2}$<br>$3.6\times10^{-2}$ | | Yaws (2003)<br>Gharagheizi et al. (2012)<br>Gharagheizi et al. (2010) | X<br>Q<br>Q | 237<br><br>246 |
| 8-methyl-1-nonanol<br>$C_{10}H_{22}O$<br>(isodecanol)<br>[25339-17-7]<br>PLLBRTOLHQQAQQ-UHFFFAOYSA-N | $1.8\times10^{-1}$ | | HSDB (2015) | Q | 99 |
| 8-methyl-2-nonanol<br>$C_{10}H_{22}O$<br>[14779-92-1]<br>ZVZKLBCGIYWGOU-UHFFFAOYSA-N | $3.2\times10^{-2}$<br>$8.7\times10^{-2}$<br>$3.4\times10^{-2}$ | | Yaws (2003)<br>Gharagheizi et al. (2012)<br>Gharagheizi et al. (2010) | X<br>Q<br>Q | 237<br><br>246 |
| 2,2-dimethyl-1-octanol<br>$C_{10}H_{22}O$<br>[2370-14-1]<br>KEXGXAGJHHCTKD-UHFFFAOYSA-N | $3.3\times10^{-2}$<br>$1.6\times10^{-2}$<br>$3.4\times10^{-2}$<br>$5.7\times10^{-1}$ | | Yaws (2003)<br>Gharagheizi et al. (2012)<br>Gharagheizi et al. (2010)<br>Yaws et al. (1997) | X<br>Q<br>Q<br>Q | 237<br><br>246 |





Table A3.2: Alcohols (ROH) (. . . continued)

| Substance Formula (Trivial Name) [CAS Registry Number] InChIKey | $H_s^{cp}$ (at $T^\ominus$) $\left[\dfrac{\mathrm{mol}}{\mathrm{m^3\,Pa}}\right]$ | $\dfrac{\mathrm{d}\ln H_s^{cp}}{\mathrm{d}(1/T)}$ [K] | Reference | Type | Note |
|---|---|---|---|---|---|
| 2,2-dimethyl-3-octanol $C_{10}H_{22}O$ [19841-72-6] AZIWNYAWODSERA-UHFFFAOYSA-N | $3.9\times10^{-2}$ $1.6\times10^{-2}$ $4.0\times10^{-2}$ | | Yaws (2003) Gharagheizi et al. (2012) Gharagheizi et al. (2010) | X Q Q | 237 246 |
| 2,2-dimethyl-4-octanol $C_{10}H_{22}O$ [66719-52-6] IVLDCRDTRQCTIY-UHFFFAOYSA-N | $4.2\times10^{-2}$ $3.0\times10^{-2}$ $3.5\times10^{-2}$ $1.0$ | | Yaws (2003) Gharagheizi et al. (2012) Gharagheizi et al. (2010) Yaws et al. (1997) | X Q Q Q | 237 246 |
| 2,3-dimethyl-3-octanol $C_{10}H_{22}O$ [19781-10-3] AWHYRPPRRQITHX-UHFFFAOYSA-N | $4.2\times10^{-2}$ $4.3\times10^{-2}$ $3.9\times10^{-2}$ $1.0$ | | Yaws (2003) Gharagheizi et al. (2012) Gharagheizi et al. (2010) Yaws et al. (1997) | X Q Q Q | 237 246 |
| 2,4-dimethyl-2-octanol $C_{10}H_{22}O$ [18675-20-2] WHJCLXRXPLJSSV-UHFFFAOYSA-N | $2.7\times10^{-2}$ $3.6\times10^{-2}$ | | Yaws (2003) Gharagheizi et al. (2010) | X Q | 237 246 |
| 2,4-dimethyl-4-octanol $C_{10}H_{22}O$ [33933-79-8] VRFMFHBPJUSGFK-UHFFFAOYSA-N | $4.1\times10^{-2}$ $6.7\times10^{-2}$ $3.7\times10^{-2}$ | | Yaws (2003) Gharagheizi et al. (2012) Gharagheizi et al. (2010) | X Q Q | 237 246 |
| 2,5-dimethyl-4-octanol $C_{10}H_{22}O$ [66719-53-7] VIHAZZPWGKHJQE-UHFFFAOYSA-N | $4.5\times10^{-2}$ $2.1\times10^{-2}$ $3.8\times10^{-2}$ | | Yaws (2003) Gharagheizi et al. (2012) Gharagheizi et al. (2010) | X Q Q | 237 246 |
| 2,6-dimethyl-1-octanol $C_{10}H_{22}O$ [62417-08-7] ZKWASVBZSAGDBI-UHFFFAOYSA-N | $3.5\times10^{-2}$ $2.4\times10^{-2}$ $3.3\times10^{-2}$ | | Yaws (2003) Gharagheizi et al. (2012) Gharagheizi et al. (2010) | X Q Q | 237 246 |
| 2,6-dimethyl-2-octanol $C_{10}H_{22}O$ [18479-57-7] WRFXXJKURVTLSY-UHFFFAOYSA-N | $2.7\times10^{-2}$ $3.6\times10^{-2}$ | | Yaws (2003) Gharagheizi et al. (2010) | X Q | 237 246 |
| 2,6-dimethyl-4-octanol $C_{10}H_{22}O$ [66719-54-8] ZFOGJEKQQNVCFB-UHFFFAOYSA-N | $3.5\times10^{-2}$ $3.5\times10^{-2}$ $3.6\times10^{-2}$ | | Yaws (2003) Gharagheizi et al. (2012) Gharagheizi et al. (2010) | X Q Q | 237 246 |
| 2,7-dimethyl-2-octanol $C_{10}H_{22}O$ [42007-73-8] JJZZVQICFJITMO-UHFFFAOYSA-N | $2.7\times10^{-2}$ $3.6\times10^{-2}$ | | Yaws (2003) Gharagheizi et al. (2010) | X Q | 237 246 |





Table A3.2: Alcohols (ROH) (. . . continued)

| Substance Formula (Trivial Name) [CAS Registry Number] InChIKey | $H_s^{cp}$ (at $T^{\ominus}$) $\left[\dfrac{\text{mol}}{\text{m}^3\,\text{Pa}}\right]$ | $\dfrac{\text{d}\ln H_s^{cp}}{\text{d}(1/T)}$ [K] | Reference | Type | Note |
|---|---|---|---|---|---|
| 2,7-dimethyl-3-octanol | $4.3\times10^{-2}$ | | Yaws (2003) | X | 237 |
| $C_{10}H_{22}O$ | $2.3\times10^{-2}$ | | Gharagheizi et al. (2012) | Q | |
| [66719-55-9] | $3.8\times10^{-2}$ | | Gharagheizi et al. (2010) | Q | 246 |
| VFUOFVONEDBLIJ-UHFFFAOYSA-N | $9.0\times10^{-1}$ | | Yaws et al. (1997) | Q | |
| 2,7-dimethyl-4-octanol | $2.3\times10^{-2}$ | | Yaws (2003) | X | 237 |
| $C_{10}H_{22}O$ | $4.2\times10^{-2}$ | | Gharagheizi et al. (2012) | Q | |
| [19781-11-4] | $3.6\times10^{-2}$ | | Gharagheizi et al. (2010) | Q | 246 |
| FRJOBNOTOHIMIH-UHFFFAOYSA-N | | | | | |
| 3,4-dimethyl-4-octanol | $4.1\times10^{-2}$ | | Yaws (2003) | X | 237 |
| $C_{10}H_{22}O$ | $4.5\times10^{-2}$ | | Gharagheizi et al. (2012) | Q | |
| [66719-30-0] | $3.9\times10^{-2}$ | | Gharagheizi et al. (2010) | Q | 246 |
| JEBZCASOCDXENQ-UHFFFAOYSA-N | | | | | |
| 3,5-dimethyl-3-octanol | $4.0\times10^{-2}$ | | Yaws (2003) | X | 237 |
| $C_{10}H_{22}O$ | $7.0\times10^{-2}$ | | Gharagheizi et al. (2012) | Q | |
| [56065-42-0] | $3.7\times10^{-2}$ | | Gharagheizi et al. (2010) | Q | 246 |
| DMIBTMSBSULJMT-UHFFFAOYSA-N | | | | | |
| 3,6-dimethyl-3-octanol | $4.0\times10^{-2}$ | | Yaws (2003) | X | 237 |
| $C_{10}H_{22}O$ | $6.9\times10^{-2}$ | | Gharagheizi et al. (2012) | Q | |
| [151-19-9] | $3.7\times10^{-2}$ | | Gharagheizi et al. (2010) | Q | 246 |
| NPHCXUPGMINOPP-UHFFFAOYSA-N | $9.6\times10^{-1}$ | | Yaws et al. (1997) | Q | |
| 3,6-dimethyl-4-octanol | $4.5\times10^{-2}$ | | Yaws (2003) | X | 237 |
| $C_{10}H_{22}O$ | $2.1\times10^{-2}$ | | Gharagheizi et al. (2012) | Q | |
| [66719-31-1] | $3.8\times10^{-2}$ | | Gharagheizi et al. (2010) | Q | 246 |
| XFUFFAQPWAGSDK-UHFFFAOYSA-N | | | | | |
| 3,7-dimethyl-1-octanol | $3.3\times10^{-2}$ | | Yaws (2003) | X | 258 |
| $C_{10}H_{22}O$ | $3.3\times10^{-2}$ | | Yaws (2003) | X | 237 |
| (pelargol) | $3.7\times10^{-1}$ | | Dupeux et al. (2022) | Q | 259 |
| [106-21-8] | $3.8\times10^{-2}$ | | Gharagheizi et al. (2012) | Q | |
| PRNCMAKCNVRZFX-UHFFFAOYSA-N | $3.2\times10^{-2}$ | | Gharagheizi et al. (2010) | Q | 246 |
| | $5.0\times10^{-1}$ | | Yaws et al. (1997) | Q | |
| 3,7-dimethyl-2-octanol | $3.0\times10^{-2}$ | | Yaws (2003) | X | 237 |
| $C_{10}H_{22}O$ | $5.6\times10^{-2}$ | | Gharagheizi et al. (2012) | Q | |
| [15340-96-2] | $3.6\times10^{-2}$ | | Gharagheizi et al. (2010) | Q | 246 |
| XCWMPEYBKUYTLZ-UHFFFAOYSA-N | | | | | |
| 3,7-dimethyl-3-octanol | $3.7\times10^{-2}$ | | Yaws (2003) | X | 237 |
| $C_{10}H_{22}O$ | $7.7\times10^{-2}$ | | Gharagheizi et al. (2012) | Q | |
| [78-69-3] | $3.7\times10^{-2}$ | | Gharagheizi et al. (2010) | Q | 246 |
| DLHQZZUEERVIGQ-UHFFFAOYSA-N | $8.5\times10^{-1}$ | | Yaws et al. (1997) | Q | |
| 4,5-dimethyl-1-octanol | $3.3\times10^{-2}$ | | Yaws (2003) | X | 237 |
| $C_{10}H_{22}O$ | $3.8\times10^{-2}$ | | Gharagheizi et al. (2012) | Q | |
| [66719-32-2] | $3.2\times10^{-2}$ | | Gharagheizi et al. (2010) | Q | 246 |
| UQKHXPAWOMZBLN-UHFFFAOYSA-N | | | | | |



Table A3.2: Alcohols (ROH) (...continued)

| Substance<br>Formula<br>(Trivial Name)<br>[CAS Registry Number]<br>InChIKey | $H_s^{cp}$<br>(at $T^{\ominus}$)<br>$\left[\dfrac{\mathrm{mol}}{\mathrm{m}^3\,\mathrm{Pa}}\right]$ | $\dfrac{\mathrm{d}\ln H_s^{cp}}{\mathrm{d}(1/T)}$<br><br>[K] | Reference | Type | Note |
|---|---|---|---|---|---|
| 4,6-dimethyl-1-octanol<br>$C_{10}H_{22}O$<br>[66719-33-3]<br>JOWDWXPTKVUHBV-UHFFFAOYSA-N | $3.3\times10^{-2}$<br>$3.8\times10^{-2}$<br>$3.2\times10^{-2}$ | | Yaws (2003)<br>Gharagheizi et al. (2012)<br>Gharagheizi et al. (2010) | X<br>Q<br>Q | 237<br><br>246 |
| 4,6-dimethyl-4-octanol<br>$C_{10}H_{22}O$<br>[56065-43-1]<br>QJVDAQVYZBSYCB-UHFFFAOYSA-N | $4.1\times10^{-2}$<br>$6.7\times10^{-2}$<br>$3.7\times10^{-2}$ | | Yaws (2003)<br>Gharagheizi et al. (2012)<br>Gharagheizi et al. (2010) | X<br>Q<br>Q | 237<br><br>246 |
| 4,7-dimethyl-1-octanol<br>$C_{10}H_{22}O$<br>[66719-34-4]<br>UWAGZVFFZJGLJU-UHFFFAOYSA-N | $3.3\times10^{-2}$<br>$3.8\times10^{-2}$<br>$3.2\times10^{-2}$ | | Yaws (2003)<br>Gharagheizi et al. (2012)<br>Gharagheizi et al. (2010) | X<br>Q<br>Q | 237<br><br>246 |
| 4,7-dimethyl-4-octanol<br>$C_{10}H_{22}O$<br>[19781-13-6]<br>OTISWHNJCUMBFI-UHFFFAOYSA-N | $4.0\times10^{-2}$<br>$6.9\times10^{-2}$<br>$3.7\times10^{-2}$<br>$9.6\times10^{-1}$ | | Yaws (2003)<br>Gharagheizi et al. (2012)<br>Gharagheizi et al. (2010)<br>Yaws et al. (1997) | X<br>Q<br>Q<br>Q | 237<br><br>246 |
| 7,7-dimethyl-1-octanol<br>$C_{10}H_{22}O$<br>[66719-35-5]<br>RYFZXYQQFYLUHM-UHFFFAOYSA-N | $3.0\times10^{-2}$<br>$3.8\times10^{-2}$<br>$3.2\times10^{-2}$ | | Yaws (2003)<br>Gharagheizi et al. (2012)<br>Gharagheizi et al. (2010) | X<br>Q<br>Q | 237<br><br>246 |
| 2-ethyl-1-octanol<br>$C_{10}H_{22}O$<br>[20592-10-3]<br>HTRVTKUOKQWGMO-UHFFFAOYSA-N | $3.9\times10^{-2}$<br>$2.3\times10^{-2}$<br>$3.6\times10^{-2}$ | | Yaws (2003)<br>Gharagheizi et al. (2012)<br>Gharagheizi et al. (2010) | X<br>Q<br>Q | 237<br><br>246 |
| 3-ethyl-1-octanol<br>$C_{10}H_{22}O$<br>[66719-36-6]<br>VXOCEIUJVCHLRR-UHFFFAOYSA-N | $3.9\times10^{-2}$<br>$3.4\times10^{-2}$<br>$3.5\times10^{-2}$ | | Yaws (2003)<br>Gharagheizi et al. (2012)<br>Gharagheizi et al. (2010) | X<br>Q<br>Q | 237<br><br>246 |
| 3-ethyl-3-octanol<br>$C_{10}H_{22}O$<br>[2051-32-3]<br>NPQPNSNHYJTUSA-UHFFFAOYSA-N | $3.9\times10^{-2}$<br>$5.7\times10^{-2}$<br>$4.3\times10^{-2}$<br>$7.7\times10^{-1}$ | | Yaws (2003)<br>Gharagheizi et al. (2012)<br>Gharagheizi et al. (2010)<br>Yaws et al. (1997) | X<br>Q<br>Q<br>Q | 237<br><br>246 |
| 3-ethyl-4-octanol<br>$C_{10}H_{22}O$<br>[63126-48-7]<br>WXJDPIWUPZLVSF-UHFFFAOYSA-N | $4.8\times10^{-2}$<br>$2.2\times10^{-2}$<br>$3.9\times10^{-2}$ | | Yaws (2003)<br>Gharagheizi et al. (2012)<br>Gharagheizi et al. (2010) | X<br>Q<br>Q | 237<br><br>246 |
| 4-ethyl-4-octanol<br>$C_{10}H_{22}O$<br>[38395-42-5]<br>OYBUBRUQIKTRET-UHFFFAOYSA-N | $4.3\times10^{-2}$<br>$4.7\times10^{-2}$<br>$4.3\times10^{-2}$ | | Yaws (2003)<br>Gharagheizi et al. (2012)<br>Gharagheizi et al. (2010) | X<br>Q<br>Q | 237<br><br>246 |





Table A3.2: Alcohols (ROH) (...continued)

| Substance<br>Formula<br>(Trivial Name)<br>[CAS Registry Number]<br>InChIKey | $H_s^{cp}$<br>(at $T^{\ominus}$)<br>$\left[\dfrac{\mathrm{mol}}{\mathrm{m^3\,Pa}}\right]$ | $\dfrac{\mathrm{d}\ln H_s^{cp}}{\mathrm{d}(1/T)}$<br><br>[K] | Reference | Type | Note |
|---|---|---|---|---|---|
| 6-ethyl-3-octanol<br>$C_{10}H_{22}O$<br>[19781-27-2]<br>JWZFCOOWAQHCBP-UHFFFAOYSA-N | $4.3\times10^{-2}$<br>$3.8\times10^{-2}$<br>$3.6\times10^{-2}$ | | Yaws (2003)<br>Gharagheizi et al. (2012)<br>Gharagheizi et al. (2010) | X<br>Q<br>Q | 237<br><br>246 |
| 2,2,3-trimethyl-3-heptanol<br>$C_{10}H_{22}O$<br>[29772-40-5]<br>GCLFPNQSLCWZAQ-UHFFFAOYSA-N | $3.7\times10^{-2}$<br>$2.7\times10^{-2}$<br>$3.6\times10^{-2}$<br>$1.1$ | | Yaws (2003)<br>Gharagheizi et al. (2012)<br>Gharagheizi et al. (2010)<br>Yaws et al. (1997) | X<br>Q<br>Q<br>Q | 237<br><br>246 |
| 2,2,4-trimethyl-4-heptanol<br>$C_{10}H_{22}O$<br>[57233-31-5]<br>KNACVESXMPHLNM-UHFFFAOYSA-N | $4.1\times10^{-2}$<br>$3.4\times10^{-2}$<br>$1.3$ | | Yaws (2003)<br>Gharagheizi et al. (2010)<br>Yaws et al. (1997) | X<br>Q<br>Q | 237<br>246 |
| 2,2,5-trimethyl-4-heptanol<br>$C_{10}H_{22}O$<br>[66256-42-6]<br>JHHGGYCQTXZCTL-UHFFFAOYSA-N | $4.4\times10^{-2}$<br>$3.6\times10^{-2}$ | | Yaws (2003)<br>Gharagheizi et al. (2010) | X<br>Q | 237<br>246 |
| 2,2,6-trimethyl-3-heptanol<br>$C_{10}H_{22}O$<br>[66256-43-7]<br>AXUPPSSNJCYJOX-UHFFFAOYSA-N | $3.8\times10^{-2}$<br>$1.4\times10^{-2}$<br>$3.8\times10^{-2}$ | | Yaws (2003)<br>Gharagheizi et al. (2012)<br>Gharagheizi et al. (2010) | X<br>Q<br>Q | 237<br><br>246 |
| 2,2,6-trimethyl-4-heptanol<br>$C_{10}H_{22}O$<br>[66256-44-8]<br>ITTQYDBLGDGLMB-UHFFFAOYSA-N | $4.4\times10^{-2}$<br>$3.4\times10^{-2}$ | | Yaws (2003)<br>Gharagheizi et al. (2010) | X<br>Q | 237<br>246 |
| 2,3,6-trimethyl-3-heptanol<br>$C_{10}H_{22}O$<br>[58046-40-5]<br>BAZNZYUOJWCTIQ-UHFFFAOYSA-N | $3.8\times10^{-2}$<br>$4.4\times10^{-2}$<br>$3.5\times10^{-2}$ | | Yaws (2003)<br>Gharagheizi et al. (2012)<br>Gharagheizi et al. (2010) | X<br>Q<br>Q | 237<br><br>246 |
| 2,4,5-trimethyl-4-heptanol<br>$C_{10}H_{22}O$<br>[66256-46-0]<br>RCSCLTAIPNDFGC-UHFFFAOYSA-N | $4.4\times10^{-2}$<br>$3.5\times10^{-2}$ | | Yaws (2003)<br>Gharagheizi et al. (2010) | X<br>Q | 237<br>246 |
| 2,4,6-trimethyl-2-heptanol<br>$C_{10}H_{22}O$<br>[66256-47-1]<br>VUZHXHGZVRACMM-UHFFFAOYSA-N | $3.6\times10^{-2}$<br>$1.0\times10^{-1}$<br>$3.3\times10^{-2}$ | | Yaws (2003)<br>Gharagheizi et al. (2012)<br>Gharagheizi et al. (2010) | X<br>Q<br>Q | 237<br><br>246 |
| 2,4,6-trimethyl-4-heptanol<br>$C_{10}H_{22}O$<br>[60836-07-9]<br>QSVYJSJPLCSACO-UHFFFAOYSA-N | $4.4\times10^{-2}$<br>$3.4\times10^{-2}$<br>$1.3$ | | Yaws (2003)<br>Gharagheizi et al. (2010)<br>Yaws et al. (1997) | X<br>Q<br>Q | 237<br>246 |



Table A3.2: Alcohols (ROH) (...continued)

| Substance<br>Formula<br>(Trivial Name)<br>[CAS Registry Number]<br>InChIKey | $H_s^{cp}$<br>(at $T^\ominus$)<br>$\left[\dfrac{\mathrm{mol}}{\mathrm{m^3\,Pa}}\right]$ | $\dfrac{\mathrm{d}\ln H_s^{cp}}{\mathrm{d}(1/T)}$<br><br>[K] | Reference | Type | Note |
|---|---|---|---|---|---|
| 2,5,6-trimethyl-2-heptanol<br>$C_{10}H_{22}O$<br>[66256-48-2]<br>LQQYOLMGDBGEBC-UHFFFAOYSA-N | $3.6\times10^{-2}$<br>$1.1\times10^{-1}$<br>$3.3\times10^{-2}$<br>$9.3\times10^{-1}$ | | Yaws (2003)<br>Gharagheizi et al. (2012)<br>Gharagheizi et al. (2010)<br>Yaws et al. (1997) | X<br>Q<br>Q<br>Q | 237<br><br>246<br> |
| 3,3,6-trimethyl-4-heptanol<br>$C_{10}H_{22}O$<br>LTDMOZOPTDNTLG-UHFFFAOYSA-N | $4.4\times10^{-2}$<br>$3.8\times10^{-2}$ | | Yaws (2003)<br>Gharagheizi et al. (2010) | X<br>Q | 237<br>246 |
| 3,5,5-trimethyl-3-heptanol<br>$C_{10}H_{22}O$<br>[66256-50-6]<br>WOUVARFKMJIYBY-UHFFFAOYSA-N | $3.1\times10^{-2}$<br>$7.7\times10^{-2}$<br>$3.4\times10^{-2}$<br>$8.6\times10^{-1}$ | | Yaws (2003)<br>Gharagheizi et al. (2012)<br>Gharagheizi et al. (2010)<br>Yaws et al. (1997) | X<br>Q<br>Q<br>Q | 237<br><br>246<br> |
| 4,6,6-trimethyl-2-heptanol<br>$C_{10}H_{22}O$<br>[51079-79-9]<br>FHQUDZUTAZYJRH-UHFFFAOYSA-N | $3.6\times10^{-2}$<br>$4.8\times10^{-2}$<br>$3.2\times10^{-2}$ | | Yaws (2003)<br>Gharagheizi et al. (2012)<br>Gharagheizi et al. (2010) | X<br>Q<br>Q | 237<br><br>246 |
| 3-(1-methylethyl)-1-heptanol<br>$C_{10}H_{22}O$<br>[38514-15-7]<br>NTGBCQFKKOCODF-UHFFFAOYSA-N | $3.1\times10^{-2}$<br>$4.2\times10^{-2}$<br>$3.2\times10^{-2}$<br>$4.4\times10^{-1}$ | | Yaws (2003)<br>Gharagheizi et al. (2012)<br>Gharagheizi et al. (2010)<br>Yaws et al. (1997) | X<br>Q<br>Q<br>Q | 237<br><br>246<br> |
| 4-(1-methylethyl)-4-heptanol<br>$C_{10}H_{22}O$<br>[51200-82-9]<br>OHSMKBPEDBXYDU-UHFFFAOYSA-N | $4.2\times10^{-2}$<br>$3.0\times10^{-2}$<br>$4.1\times10^{-2}$<br>$1.0$ | | Yaws (2003)<br>Gharagheizi et al. (2012)<br>Gharagheizi et al. (2010)<br>Yaws et al. (1997) | X<br>Q<br>Q<br>Q | 237<br><br>246<br> |
| 3-ethyl-2-methyl-3-heptanol<br>$C_{10}H_{22}O$<br>[66719-37-7]<br>GYCRUUSYQLIIBA-UHFFFAOYSA-N | $4.0\times10^{-2}$<br>$3.3\times10^{-2}$<br>$4.1\times10^{-2}$<br>$9.3\times10^{-1}$ | | Yaws (2003)<br>Gharagheizi et al. (2012)<br>Gharagheizi et al. (2010)<br>Yaws et al. (1997) | X<br>Q<br>Q<br>Q | 237<br><br>246<br> |
| 3-ethyl-5-methyl-3-heptanol<br>$C_{10}H_{22}O$<br>FGJMKJVRSBVZPS-UHFFFAOYSA-N | $4.0\times10^{-2}$<br>$4.8\times10^{-2}$<br>$3.9\times10^{-2}$ | | Yaws (2003)<br>Gharagheizi et al. (2012)<br>Gharagheizi et al. (2010) | X<br>Q<br>Q | 237<br><br>246 |
| 5-ethyl-4-methyl-3-heptanol<br>$C_{10}H_{22}O$<br>[66731-94-0]<br>XVSHXHABYNUHNI-UHFFFAOYSA-N | $4.4\times10^{-2}$<br>$2.2\times10^{-2}$<br>$3.8\times10^{-2}$ | | Yaws (2003)<br>Gharagheizi et al. (2012)<br>Gharagheizi et al. (2010) | X<br>Q<br>Q | 237<br><br>246 |
| 2-propyl-1-heptanol<br>$C_{10}H_{22}O$<br>[10042-59-8]<br>YLQLIQIAXYRMDL-UHFFFAOYSA-N | $3.3\times10^{-2}$<br>$3.0\times10^{-2}$<br>$3.6\times10^{-2}$<br>$4.0\times10^{-1}$ | | Yaws (2003)<br>Gharagheizi et al. (2012)<br>Gharagheizi et al. (2010)<br>Yaws et al. (1997) | X<br>Q<br>Q<br>Q | 237<br><br>246<br> |
| 4-propyl-4-heptanol<br>$C_{10}H_{22}O$<br>[2198-72-3]<br>SJTPBRMACCDJPZ-UHFFFAOYSA-N | $4.2\times10^{-2}$<br>$4.9\times10^{-2}$<br>$4.3\times10^{-2}$<br>$9.0\times10^{-1}$ | | Yaws (2003)<br>Gharagheizi et al. (2012)<br>Gharagheizi et al. (2010)<br>Yaws et al. (1997) | X<br>Q<br>Q<br>Q | 237<br><br>246<br> |



Table A3.2: Alcohols (ROH) (...continued)

| Substance Formula (Trivial Name) [CAS Registry Number] InChIKey | $H_s^{cp}$ (at $T^{\ominus}$) $\left[\dfrac{\text{mol}}{\text{m}^3\,\text{Pa}}\right]$ | $\dfrac{\text{d}\ln H_s^{cp}}{\text{d}(1/T)}$ [K] | Reference | Type | Note |
|---|---|---|---|---|---|
| 2,2,3,4-tetramethyl-3-hexanol $C_{10}H_{22}O$ [66256-63-1] SAFMWGCEQYLRLS-UHFFFAOYSA-N | $3.1\times10^{-2}$ $2.2\times10^{-2}$ $3.3\times10^{-2}$ $9.6\times10^{-1}$ | | Yaws (2003) Gharagheizi et al. (2012) Gharagheizi et al. (2010) Yaws et al. (1997) | X Q Q Q | 237 246 |
| 2,2,4,4-tetramethyl-3-hexanol $C_{10}H_{22}O$ [66256-65-3] LKAPNEGVXBMYKE-UHFFFAOYSA-N | $3.2\times10^{-2}$ $6.7\times10^{-3}$ $4.0\times10^{-2}$ $1.0$ | | Yaws (2003) Gharagheizi et al. (2012) Gharagheizi et al. (2010) Yaws et al. (1997) | X Q Q Q | 237 246 |
| 2,2,5,5-tetramethyl-3-hexanol $C_{10}H_{22}O$ [55073-86-4] CFEYPBVKHMZCFR-UHFFFAOYSA-N | $4.6\times10^{-2}$ $3.6\times10^{-2}$ $1.7$ | | Yaws (2003) Gharagheizi et al. (2010) Yaws et al. (1997) | X Q Q | 237 246 |
| 2,3,4,4-tetramethyl-2-hexanol $C_{10}H_{22}O$ [66256-66-4] IELWGTFQRBDDCZ-UHFFFAOYSA-N | $3.2\times10^{-2}$ $6.7\times10^{-2}$ $3.0\times10^{-2}$ $1.0$ | | Yaws (2003) Gharagheizi et al. (2012) Gharagheizi et al. (2010) Yaws et al. (1997) | X Q Q Q | 237 246 |
| 2,3,4,4-tetramethyl-3-hexanol $C_{10}H_{22}O$ [66256-67-5] ZQFIWHOYWCJIHU-UHFFFAOYSA-N | $2.5\times10^{-2}$ $2.9\times10^{-2}$ $3.3\times10^{-2}$ $7.3\times10^{-1}$ | | Yaws (2003) Gharagheizi et al. (2012) Gharagheizi et al. (2010) Yaws et al. (1997) | X Q Q Q | 237 246 |
| 2,3,5,5-tetramethyl-3-hexanol $C_{10}H_{22}O$ [5396-09-8] KTSYBVKVNWGHBC-UHFFFAOYSA-N | $2.5\times10^{-2}$ $6.4\times10^{-2}$ $3.1\times10^{-2}$ | | Yaws (2003) Gharagheizi et al. (2012) Gharagheizi et al. (2010) | X Q Q | 237 246 |
| 2,4,4,5-tetramethyl-3-hexanol $C_{10}H_{22}O$ [66256-68-6] VVRHXTVKOHXZJX-UHFFFAOYSA-N | $2.8\times10^{-2}$ $1.4\times10^{-2}$ $3.8\times10^{-2}$ | | Yaws (2003) Gharagheizi et al. (2012) Gharagheizi et al. (2010) | X Q Q | 237 246 |
| 3,3,5,5-tetramethyl-2-hexanol $C_{10}H_{22}O$ [66256-69-7] USWOQOMRDCYQMW-UHFFFAOYSA-N | $3.2\times10^{-2}$ $2.1\times10^{-2}$ $3.4\times10^{-2}$ | | Yaws (2003) Gharagheizi et al. (2012) Gharagheizi et al. (2010) | X Q Q | 237 246 |
| 3,4,4,5-tetramethyl-3-hexanol $C_{10}H_{22}O$ [66256-39-1] UHJJSHWWLQIIOU-UHFFFAOYSA-N | $2.5\times10^{-2}$ $4.4\times10^{-2}$ $3.2\times10^{-2}$ $7.0\times10^{-1}$ | | Yaws (2003) Gharagheizi et al. (2012) Gharagheizi et al. (2010) Yaws et al. (1997) | X Q Q Q | 237 246 |
| 3,4,5,5-tetramethyl-3-hexanol $C_{10}H_{22}O$ [66256-40-4] LBLUJHQRJVUQJR-UHFFFAOYSA-N | $2.9\times10^{-2}$ $5.2\times10^{-2}$ $3.1\times10^{-2}$ $8.8\times10^{-1}$ | | Yaws (2003) Gharagheizi et al. (2012) Gharagheizi et al. (2010) Yaws et al. (1997) | X Q Q Q | 237 246 |





Table A3.2: Alcohols (ROH) (. . . continued)

| Substance / Formula / (Trivial Name) / [CAS Registry Number] / InChIKey | $H_s^{cp}$ (at $T^{\ominus}$) $\left[\dfrac{\text{mol}}{\text{m}^3\,\text{Pa}}\right]$ | $\dfrac{\text{d}\ln H_s^{cp}}{\text{d}(1/T)}$ [K] | Reference | Type | Note |
|---|---|---|---|---|---|
| 2,4-dimethyl-4-ethyl-3-hexanol C$_{10}$H$_{22}$O [66719-48-0] NLGGBBOPZSANIC-UHFFFAOYSA-N | $4.0\times10^{-2}$ $4.0\times10^{-2}$ | | Yaws (2003) Gharagheizi et al. (2010) | X Q | 237 246 |
| 2-butyl-1-hexanol C$_{10}$H$_{22}$O [2768-15-2] LAPPDPWPIZBBJY-UHFFFAOYSA-N | $4.4\times10^{-1}$ | | Yaws et al. (1997) | Q | |
| 4-ethyl-2,2-dimethyl-3-hexanol C$_{10}$H$_{22}$O [66719-47-9] UOYRABPBVGDJLW-UHFFFAOYSA-N | $4.0\times10^{-2}$ $4.0\times10^{-2}$ $1.1$ | | Yaws (2003) Gharagheizi et al. (2010) Yaws et al. (1997) | X Q Q | 237 246 |
| 4-methyl-2-(1-methylethyl)-1-hexanol C$_{10}$H$_{22}$O [66719-41-3] VHGVOODVCUKVCL-UHFFFAOYSA-N | $3.7\times10^{-2}$ $1.8\times10^{-2}$ $3.0\times10^{-2}$ $7.3\times10^{-1}$ | | Yaws (2003) Gharagheizi et al. (2012) Gharagheizi et al. (2010) Yaws et al. (1997) | X Q Q Q | 237 246 |
| 4-methyl-2-propyl-1-hexanol C$_{10}$H$_{22}$O [66256-62-0] VZXWJVFQXZUFQS-UHFFFAOYSA-N | $3.6\times10^{-2}$ $2.3\times10^{-2}$ $3.3\times10^{-2}$ $5.7\times10^{-1}$ | | Yaws (2003) Gharagheizi et al. (2012) Gharagheizi et al. (2010) Yaws et al. (1997) | X Q Q Q | 237 246 |
| 5,5-dimethyl-2-ethyl-1-hexanol C$_{10}$H$_{22}$O DXDJMJLAABGHOH-UHFFFAOYSA-N | $2.7\times10^{-2}$ $2.7\times10^{-2}$ $3.1\times10^{-2}$ | | Yaws (2003) Gharagheizi et al. (2012) Gharagheizi et al. (2010) | X Q Q | 237 246 |
| 5,5-dimethyl-3-ethyl-3-hexanol C$_{10}$H$_{22}$O [5340-62-5] FVVBGHIXYRKNGT-UHFFFAOYSA-N | $3.7\times10^{-2}$ $4.1\times10^{-2}$ $3.5\times10^{-2}$ | | Yaws (2003) Gharagheizi et al. (2012) Gharagheizi et al. (2010) | X Q Q | 237 246 |
| 5-methyl-2-(1-methylethyl)-1-hexanol C$_{10}$H$_{22}$O [2051-33-4] SFIQHFBITUEIBP-UHFFFAOYSA-N | $3.0\times10^{-2}$ $2.7\times10^{-2}$ $3.0\times10^{-2}$ $4.8\times10^{-1}$ | | Yaws (2003) Gharagheizi et al. (2012) Gharagheizi et al. (2010) Yaws et al. (1997) | X Q Q Q | 237 246 |
| 2-methyl-3-(1-methylethyl)-3-hexanol C$_{10}$H$_{22}$O (2,4-dimethyl-3-propyl-3-pentanol) [51200-81-8] GGOIYPAZTVTJBE-UHFFFAOYSA-N | $4.3\times10^{-2}$ $3.7\times10^{-2}$ $2.1\times10^{-2}$ $1.7\times10^{-2}$ $3.7\times10^{-2}$ $3.7\times10^{-2}$ $9.6\times10^{-1}$ $1.2$ | | Yaws (2003) Yaws (2003) Gharagheizi et al. (2012) Gharagheizi et al. (2012) Gharagheizi et al. (2010) Gharagheizi et al. (2010) Yaws et al. (1997) Yaws et al. (1997) | X X Q Q Q Q Q Q | 237 237 246 246 |





Table A3.2: Alcohols (ROH) (...continued)

| Substance<br>Formula<br>(Trivial Name)<br>[CAS Registry Number]<br>InChIKey | $H_s^{cp}$<br>(at $T^\ominus$)<br>$\left[\dfrac{\mathrm{mol}}{\mathrm{m^3\,Pa}}\right]$ | $\dfrac{\mathrm{d}\ln H_s^{cp}}{\mathrm{d}(1/T)}$<br><br>[K] | Reference | Type | Note |
|---|---|---|---|---|---|
| 2,2,3,4,4-pentamethyl-3-pentanol | $2.4\times10^{-2}$ | | Yaws (2003) | X | 237 |
| $C_{10}H_{22}O$ | $1.7\times10^{-2}$ | | Gharagheizi et al. (2012) | Q | |
| [5857-69-2] | $3.1\times10^{-2}$ | | Gharagheizi et al. (2010) | Q | 246 |
| UYFZQUABCZILAI-UHFFFAOYSA-N | $8.9\times10^{-1}$ | | Yaws et al. (1997) | Q | |
| 2,3-dimethyl-2-*tert*-butyl-1-butanol | $3.0\times10^{-2}$ | | Yaws (2003) | X | 237 |
| $C_{10}H_{22}O$ | $1.1\times10^{-2}$ | | Gharagheizi et al. (2012) | Q | |
| [81931-81-9] | $3.1\times10^{-2}$ | | Gharagheizi et al. (2010) | Q | 246 |
| WQRJEFGWAMKEBO-UHFFFAOYSA-N | | | | | |
| 2,4-dimethyl-3-(1-methylethyl)-3-pentanol | $3.2\times10^{-2}$ | | Yaws (2003) | X | 237 |
| $C_{10}H_{22}O$ | $1.6\times10^{-2}$ | | Gharagheizi et al. (2012) | Q | |
| [51200-83-0] | $3.3\times10^{-2}$ | | Gharagheizi et al. (2010) | Q | 246 |
| IFXNWIUSZUHIDG-UHFFFAOYSA-N | $8.9\times10^{-1}$ | | Yaws et al. (1997) | Q | |
| 3,4-dimethyl-3-isopropyl-2-pentanol | $4.0\times10^{-2}$ | | Yaws (2003) | X | 237 |
| $C_{10}H_{22}O$ | $3.4\times10^{-2}$ | | Gharagheizi et al. (2010) | Q | 246 |
| [66719-50-4] | | | | | |
| DNBJFVXHMLZKHP-UHFFFAOYSA-N | | | | | |
| 4,4-dimethyl-3-isopropyl-1-pentanol | $4.3\times10^{-2}$ | | Yaws (2003) | X | 237 |
| $C_{10}H_{22}O$ | $3.0\times10^{-2}$ | | Gharagheizi et al. (2010) | Q | 246 |
| [66719-51-5] | | | | | |
| DQWFOEIWLQBIFY-UHFFFAOYSA-N | | | | | |
| 3-ethyl-2,2,4-trimethyl-3-pentanol | $3.1\times10^{-2}$ | | Yaws (2003) | X | 237 |
| $C_{10}H_{22}O$ | $1.5\times10^{-2}$ | | Gharagheizi et al. (2012) | Q | |
| [66256-41-5] | $3.4\times10^{-2}$ | | Gharagheizi et al. (2010) | Q | 246 |
| CHZVHXPAKQDRSE-UHFFFAOYSA-N | $9.9\times10^{-1}$ | | Yaws et al. (1997) | Q | |
| 4-methyl-2-(2-methylpropyl)-1-pentanol | $3.5\times10^{-2}$ | | Yaws (2003) | X | 237 |
| $C_{10}H_{22}O$ | $2.0\times10^{-2}$ | | Gharagheizi et al. (2012) | Q | |
| [22417-45-4] | $3.0\times10^{-2}$ | | Gharagheizi et al. (2010) | Q | 246 |
| FXXCTCCAZGTNNO-UHFFFAOYSA-N | $6.6\times10^{-1}$ | | Yaws et al. (1997) | Q | |
| 1-undecanol | $1.4\times10^{-1}$ | | Brockbank (2013) | L | |
| $C_{11}H_{24}O$ | $2.2\times10^{-1}$ | | Yaws (2003) | X | 258 |
| [112-42-5] | $2.1\times10^{-1}$ | | Dupeux et al. (2022) | Q | 259 |
| KJIOQYGWTQBHNH-UHFFFAOYSA-N | $1.4\times10^{-1}$ | | HSDB (2015) | Q | 99 |
| | $1.2\times10^{-1}$ | | Hilal et al. (2008) | Q | |
| | $1.7\times10^{-1}$ | | Yao et al. (2002) | Q | 229 |
| | $2.2\times10^{-1}$ | | Yaws et al. (1997) | Q | |
| | $3.0\times10^{-1}$ | | Yaws (1999) | ? | 21 |



Table A3.2: Alcohols (ROH) (...continued)

| Substance Formula (Trivial Name) [CAS Registry Number] InChIKey | $H_s^{cp}$ (at $T^\ominus$) $\left[\dfrac{\text{mol}}{\text{m}^3\,\text{Pa}}\right]$ | $\dfrac{\text{d}\ln H_s^{cp}}{\text{d}(1/T)}$ [K] | Reference | Type | Note |
|---|---|---|---|---|---|
| 2-undecanol $C_{11}H_{24}O$ [1653-30-1] XMUJIPOFTAHSOK-UHFFFAOYSA-N | $1.2\times10^{-1}$ $2.3\times10^{-1}$ | | Yaws (2003) Dupeux et al. (2022) | X Q | 258 259 |
| 3-undecanol $C_{11}H_{24}O$ [6929-08-4] HCARCYFXWDRVBZ-UHFFFAOYSA-N | $6.0\times10^{-2}$ $1.7\times10^{-1}$ $1.6\times10^{-1}$ | | Wang et al. (2017) Wang et al. (2017) Wang et al. (2017) | Q Q Q | 80, 238 80, 239 80, 240 |
| 1-dodecanol $C_{12}H_{26}O$ [112-53-8] LQZZUXJYWNFBMV-UHFFFAOYSA-N | $1.2\times10^{-1}$ $4.4\times10^{-1}$ $1.4\times10^{-1}$ $1.9\times10^{-1}$ $1.8\times10^{-1}$ $2.4\times10^{-1}$ $1.5$ $7.8\times10^{-2}$ $9.9\times10^{-2}$ $9.9\times10^{-2}$ $1.5\times10^{-1}$ $3.5\times10^{-1}$ $2.1\times10^{-1}$ $1.6\times10^{-1}$ $1.9\times10^{-1}$ $4.4\times10^{-1}$ $1.9\times10^{-1}$ $1.1\times10^{-1}$ | 9800 | Brockbank (2013) Altschuh et al. (1999) Abraham (1984) Yaws (2003) Dupeux et al. (2022) Keshavarz et al. (2022) Duchowicz et al. (2020) Raventos-Duran et al. (2010) Raventos-Duran et al. (2010) Raventos-Duran et al. (2010) Hilal et al. (2008) Modarresi et al. (2007) Yaffe et al. (2003) Yao et al. (2002) Yaws et al. (1997) Duchowicz et al. (2020) Yaws (1999) Yaws and Yang (1992) | L M V X Q Q Q Q Q Q Q Q Q Q Q ? ? ? | 258 259 184 242, 243 244 245 67 248, 249 229, 267 185, 21 21 21 |
| 3-dodecanol $C_{12}H_{26}O$ [10203-30-2] OKDGZLITBCRLLJ-UHFFFAOYSA-N | $4.8\times10^{-2}$ $1.5\times10^{-1}$ $1.5\times10^{-1}$ | | Wang et al. (2017) Wang et al. (2017) Wang et al. (2017) | Q Q Q | 80, 238 80, 239 80, 240 |
| 2,6,8-trimethyl-4-nonanol $C_{12}H_{26}O$ [123-17-1] LFEHSRSSAGQWNI-UHFFFAOYSA-N | $1.0\times10^{-1}$ $1.1\times10^{-1}$ $6.0\times10^{-2}$ $2.6\times10^{-2}$ | | Zhang et al. (2010) Zhang et al. (2010) Zhang et al. (2010) Zhang et al. (2010) | Q Q Q Q | 287, 288 287, 289 287, 290 287, 291 |
| 1-tridecanol $C_{13}H_{28}O$ [112-70-9] XFRVVPUIAFSTFO-UHFFFAOYSA-N | $1.2\times10^{-1}$ $1.5\times10^{-1}$ $7.6\times10^{-2}$ $1.3\times10^{-1}$ $1.2\times10^{-1}$ $1.8\times10^{-1}$ | | Yaws (2003) Dupeux et al. (2022) HSDB (2015) Yao et al. (2002) Yaws et al. (1997) Yaws (1999) | X Q Q Q Q ? | 258 259 99 229 21 |



Table A3.2: Alcohols (ROH) (...continued)

| Substance Formula (Trivial Name) [CAS Registry Number] InChIKey | $H_s^{cp}$ (at $T^{\ominus}$) $\left[\dfrac{\text{mol}}{\text{m}^3\,\text{Pa}}\right]$ | $\dfrac{\text{d}\ln H_s^{cp}}{\text{d}(1/T)}$ [K] | Reference | Type | Note |
|---|---|---|---|---|---|
| 1-tetradecanol | $6.2\times10^{-2}$ | | Duchowicz et al. (2020) | V | 186 |
| $C_{14}H_{30}O$ | $6.2\times10^{-2}$ | | HSDB (2015) | V | |
| [112-72-1] | $2.2\times10^{-1}$ | | Abraham (1984) | R | |
| HLZKNKRTKFSKGZ-UHFFFAOYSA-N | $9.4\times10^{-2}$ | | Yaws (2003) | X | 258 |
| | $1.2\times10^{-1}$ | | Dupeux et al. (2022) | Q | 259 |
| | 1.5 | | Duchowicz et al. (2020) | Q | |
| | $6.2\times10^{-2}$ | | Hilal et al. (2008) | Q | |
| | $2.7\times10^{-1}$ | | Modarresi et al. (2007) | Q | 67 |
| | $9.7\times10^{-2}$ | | Yaffe et al. (2003) | Q | 248, 249 |
| | $6.6\times10^{-2}$ | | Yao et al. (2002) | Q | 229 |
| | $9.5\times10^{-2}$ | | Yaws et al. (1997) | Q | |
| | $9.5\times10^{-2}$ | | Yaws (1999) | ? | 21 |
| | $3.9\times10^{3}$ | | Yaws and Yang (1992) | ? | 21, 409 |
| 2-tetradecanol | $1.7\times10^{-1}$ | | Gharagheizi et al. (2012) | Q | |
| $C_{14}H_{30}O$ | | | | | |
| [4706-81-4] | | | | | |
| BRGJIIMZXMWMCC-UHFFFAOYSA-N | | | | | |
| 1-pentadecanol | $2.2\times10^{-1}$ | | Abraham (1984) | V | |
| $C_{15}H_{32}O$ | $2.5\times10^{-2}$ | | Yaws (2003) | X | 237 |
| [629-76-5] | $8.9\times10^{-2}$ | | Gharagheizi et al. (2012) | Q | |
| REIUXOLGHVXAEO-UHFFFAOYSA-N | $2.7\times10^{-2}$ | | Gharagheizi et al. (2010) | Q | 246 |
| | $2.5\times10^{-1}$ | | Yaffe et al. (2003) | Q | 248, 249 |
| | $2.5\times10^{-1}$ | | Yaws et al. (1997) | Q | |
| | $3.0\times10^{3}$ | | Yaws and Yang (1992) | ? | 21, 410 |
| 1-hexadecanol | $2.1\times10^{-1}$ | | Duchowicz et al. (2020) | V | 186 |
| $C_{16}H_{34}O$ | $2.1\times10^{-1}$ | | HSDB (2015) | V | |
| (cetyl alcohol) | $3.5\times10^{-1}$ | | Abraham (1984) | R | |
| [124-29-8] | $6.4\times10^{-2}$ | | Yaws (2003) | X | 237 |
| BXWNKGSJHAJOGX-UHFFFAOYSA-N | 1.5 | | Duchowicz et al. (2020) | Q | |
| | $7.4\times10^{-2}$ | | Gharagheizi et al. (2012) | Q | |
| | $3.8\times10^{-2}$ | | Gharagheizi et al. (2010) | Q | 246 |
| | $3.9\times10^{-2}$ | | Hilal et al. (2008) | Q | |
| | $2.1\times10^{-1}$ | | Modarresi et al. (2007) | Q | 67 |
| | $2.5\times10^{-1}$ | | Yaffe et al. (2003) | Q | 248, 272 |
| | $1.0\times10^{-1}$ | | Yaws et al. (1997) | Q | |
| | $1.0\times10^{-1}$ | | Yaws (1999) | ? | 21 |
| | $5.9\times10^{-1}$ | | Yaws and Yang (1992) | ? | 21 |
| 2-hexadecanol | $1.6\times10^{-2}$ | | Yaws (2003) | X | 237 |
| $C_{16}H_{34}O$ | $1.1\times10^{-2}$ | | Gharagheizi et al. (2010) | Q | 246 |
| [14852-31-4] | | | | | |
| FVDRFBGMOWJEOR-UHFFFAOYSA-N | | | | | |



Table A3.2: Alcohols (ROH) (...continued)

| Substance<br>Formula<br>(Trivial Name)<br>[CAS Registry Number]<br>InChIKey | $H_s^{cp}$<br>(at $T^{\ominus}$)<br>$\left[\dfrac{\mathrm{mol}}{\mathrm{m^3\,Pa}}\right]$ | $\dfrac{\mathrm{d}\ln H_s^{cp}}{\mathrm{d}(1/T)}$<br><br>[K] | Reference | Type | Note |
|---|---|---|---|---|---|
| 1-heptadecanol<br>$C_{17}H_{36}O$<br>[1454-85-9]<br>GOQYKNQRPGWPLP-UHFFFAOYSA-N | $4.5\times10^{-2}$<br>$7.1\times10^{-2}$<br>$6.6\times10^{-2}$<br>$4.6\times10^{-2}$<br>$4.5\times10^{-2}$<br>$1.2\times10^{1}$ | | Yaws (2003)<br>Gharagheizi et al. (2012)<br>Gharagheizi et al. (2010)<br>Yaffe et al. (2003)<br>Yaws et al. (1997)<br>Yaws and Yang (1992) | X<br>Q<br>Q<br>Q<br>Q<br>? | 237<br><br>246<br>248, 249<br><br>21 |
| 2-heptadecanol<br>$C_{17}H_{36}O$<br>[16813-18-6]<br>ZNYQHFLBAPNPRC-UHFFFAOYSA-N | $8.7\times10^{-3}$<br>$1.2\times10^{-2}$ | | Yaws (2003)<br>Gharagheizi et al. (2010) | X<br>Q | 237<br>246 |
| 1-octadecanol<br>$C_{18}H_{38}O$<br>[112-92-5]<br>GLDOVTGHNKAZLK-UHFFFAOYSA-N | $1.2\times10^{-2}$<br>$1.2\times10^{-2}$<br>$3.8\times10^{-1}$<br>$1.5$<br>$2.5\times10^{-2}$<br>$1.7\times10^{-1}$<br>$3.1\times10^{-3}$<br>$3.1\times10^{-3}$<br>$1.2\times10^{-2}$<br>$9.1\times10^{-1}$ | | Duchowicz et al. (2020)<br>HSDB (2015)<br>Abraham (1984)<br>Duchowicz et al. (2020)<br>Hilal et al. (2008)<br>Modarresi et al. (2007)<br>Yaffe et al. (2003)<br>Yaws et al. (1997)<br>Yaws (1999)<br>Yaws and Yang (1992) | V<br>V<br>R<br>Q<br>Q<br>Q<br>Q<br>Q<br>?<br>? | 186<br><br><br><br><br>67<br>248, 249<br><br>21, 411<br>21, 411 |
| 1-nonadecanol<br>$C_{19}H_{40}O$<br>[1454-84-8]<br>XGFDHKJUZCCPKQ-UHFFFAOYSA-N | $9.9\times10^{-2}$ | | Yaws et al. (1997) | Q | |
| 1-eicosanol<br>$C_{20}H_{42}O$<br>[629-96-9]<br>BTFJIXJJCSYFAL-UHFFFAOYSA-N | $4.7\times10^{-1}$<br>$5.5\times10^{-2}$<br>$1.8\times10^{-2}$ | | HSDB (2015)<br>Gharagheizi et al. (2012)<br>Yaws et al. (1997) | Q<br>Q<br>Q | 99 |
| 1-docosanol<br>$C_{22}H_{46}O$<br>(behenic alcohol)<br>[661-19-8]<br>NOPFSRXAKWQILS-UHFFFAOYSA-N | $6.2\times10^{-3}$ | | HSDB (2015) | Q | 99 |
| 1-tetracosanol<br>$C_{24}H_{50}O$<br>[506-51-4]<br>TYWMIZZBOVGFOV-UHFFFAOYSA-N | $3.4\times10^{-3}$ | | HSDB (2015) | Q | 99 |
| cyclopentanol<br>$C_5H_9OH$<br>[96-41-3]<br>XCIXKGXIYUWCLL-UHFFFAOYSA-N | $4.2$<br>$2.2$<br>$4.3$<br>$3.8$<br>$2.0$<br><br>$4.4$ | $8200$<br>$5900$<br>$8000$<br><br><br>$7200$ | Plyasunov and Shock (2000)<br>Hovorka et al. (2002)<br>Cabani et al. (1975b)<br>HSDB (2015)<br>Hilal et al. (2008)<br>Kühne et al. (2005)<br>Nirmalakhandan et al. (1997) | L<br>M<br>T<br>Q<br>Q<br>Q<br>Q | <br>11<br><br>99 |



Table A3.2: Alcohols (ROH) (...continued)

| Substance<br>Formula<br>(Trivial Name)<br>[CAS Registry Number]<br>InChIKey | $H_s^{cp}$ (at $T^{\ominus}$) $\left[\dfrac{\text{mol}}{\text{m}^3\,\text{Pa}}\right]$ | $\dfrac{\text{d}\ln H_s^{cp}}{\text{d}(1/T)}$ [K] | Reference | Type | Note |
|---|---|---|---|---|---|
| | | 7300 | Kühne et al. (2005) | ? | |
| | 4.3 | | Abraham et al. (1990) | ? | |
| cyclohexanol | 4.3 | 7700 | Brockbank (2013) | L | 1 |
| $C_6H_{11}OH$ | 4.4 | 8500 | Plyasunov and Shock (2000) | L | |
| [108-93-0] | 2.5 | | Chao et al. (2017) | M | |
| HPXRVTGHNJAIIH-UHFFFAOYSA-N | 4.3 | 7700 | Hovorka et al. (2002) | M | 11 |
| | 2.2 | | Altschuh et al. (1999) | M | |
| | 2.7 | | Chao et al. (2017) | V | |
| | 4.5 | | Mackay et al. (2006c) | V | |
| | 4.5 | | Mackay et al. (1995) | V | |
| | 3.5 | | Meylan and Howard (1991) | V | |
| | 1.7 | | Hine and Mookerjee (1975) | V | |
| | 4.1 | 8500 | Cabani et al. (1975b) | T | |
| | 3.9 | | Yaws (2003) | X | 258 |
| | 3.6 | | Howard (1993) | X | 412 |
| | 6.2 | | Dupeux et al. (2022) | Q | 259 |
| | 1.1 | | Keshavarz et al. (2022) | Q | |
| | 4.1 | | Duchowicz et al. (2020) | Q | |
| | $4.5\times10^{-1}$ | | Wang et al. (2017) | Q | 80, 238 |
| | 4.5 | | Wang et al. (2017) | Q | 80, 239 |
| | 6.3 | | Wang et al. (2017) | Q | 80, 240 |
| | 1.6 | | Li et al. (2014) | Q | 241 |
| | 2.0 | | Raventos-Duran et al. (2010) | Q | 242, 243 |
| | 2.5 | | Raventos-Duran et al. (2010) | Q | 244 |
| | 2.0 | | Raventos-Duran et al. (2010) | Q | 245 |
| | 3.3 | | Hilal et al. (2008) | Q | |
| | 2.3 | | Modarresi et al. (2007) | Q | 67 |
| | | 7500 | Kühne et al. (2005) | Q | |
| | 3.8 | | Yaffe et al. (2003) | Q | 248, 249 |
| | 1.3 | | Yao et al. (2002) | Q | 229 |
| | 4.1 | | English and Carroll (2001) | Q | 230, 231 |
| | 2.3 | | Katritzky et al. (1998) | Q | |
| | 2.7 | | Nirmalakhandan et al. (1997) | Q | |
| | 3.6 | | Russell et al. (1992) | Q | 279 |
| | 2.0 | | Suzuki et al. (1992) | Q | 232 |
| | 2.0 | | Meylan and Howard (1991) | Q | |
| | 3.4 | | Nirmalakhandan and Speece (1988) | Q | |
| | 2.2 | | Duchowicz et al. (2020) | ? | 185, 21 |
| | | 7500 | Kühne et al. (2005) | ? | |
| | 3.9 | | Yaws (1999) | ? | 21 |
| | 4.1 | | Abraham et al. (1990) | ? | |
| cycloheptanol | 4.6 | 9000 | Plyasunov and Shock (2000) | L | |
| $C_7H_{13}OH$ | 4.2 | 9000 | Cabani et al. (1975b) | T | |
| [502-41-0] | 1.0 | | Hilal et al. (2008) | Q | |
| QCRFMSUKWRQZEM-UHFFFAOYSA-N | 3.7 | | English and Carroll (2001) | Q | 230, 231 |
| | 4.2 | | Abraham et al. (1990) | ? | |



Table A3.2: Alcohols (ROH) (...continued)

| Substance Formula (Trivial Name) [CAS Registry Number] InChIKey | $H_s^{cp}$ (at $T^\ominus$) $\left[\dfrac{\text{mol}}{\text{m}^3\,\text{Pa}}\right]$ | $\dfrac{\text{d}\ln H_s^{cp}}{\text{d}(1/T)}$ [K] | Reference | Type | Note |
|---|---|---|---|---|---|
| 4-methylcyclohexanol $C_7H_{13}OH$ [589-91-3] MQWCXKGKQLNYQG-UHFFFAOYSA-N | 2.5 | | Ebert et al. (2023) | ? | 318 |
| 2-methylcyclohexanol $C_7H_{14}O$ [583-59-5] NDVWOBYBJYUSMF-UHFFFAOYSA-N | 1.4 1.3 5.9 1.7 1.6 1.6 1.6 1.3 3.2 1.3 | | Chao et al. (2017) Altschuh et al. (1999) Keshavarz et al. (2022) Duchowicz et al. (2020) Raventos-Duran et al. (2010) Raventos-Duran et al. (2010) Raventos-Duran et al. (2010) Hilal et al. (2008) Modarresi et al. (2007) Duchowicz et al. (2020) | M M Q Q Q Q Q Q Q ? | 184 242, 243 244 245 67 185, 21 |
| 3-methylcyclohexanol $C_7H_{14}O$ [591-23-1] HTSABYAWKQAHBT-UHFFFAOYSA-N | 2.9 2.7 5.9 1.7 1.6 1.6 1.6 1.7 2.7 | | Chao et al. (2017) Altschuh et al. (1999) Keshavarz et al. (2022) Duchowicz et al. (2020) Raventos-Duran et al. (2010) Raventos-Duran et al. (2010) Raventos-Duran et al. (2010) Modarresi et al. (2007) Duchowicz et al. (2020) | M M Q Q Q Q Q Q ? | 299 242, 243 244 245 67 185, 21 |
| MCM:C8BCOH $C_8H_{14}O$ CEOBYJRLFASWKF-UHFFFAOYSA-N | $8.7\times10^{-1}$ 2.4 1.5 | | Wang et al. (2017) Wang et al. (2017) Wang et al. (2017) | Q Q Q | 80, 238 80, 239 80, 240 |
| (4-methylcyclohexyl)methanol $C_8H_{16}O$ [34885-03-5] OSINZLLLLCUKJH-UHFFFAOYSA-N | 1.5 | | HSDB (2015) | Q | 99 |
| borneol $C_{10}H_{18}O$ [507-70-0] DTGKSKDOIYIVQL-UHFFFAOYSA-N | $4.4\times10^{-1}$ | | Ebert et al. (2023) | ? | 316 |
| fenchol $C_{10}H_{18}O$ (fenchyl alcohol) [1632-73-1] IAIHUHQCLTYTSF-UHFFFAOYSA-N | $3.5\times10^{-1}$ | | Ebert et al. (2023) | ? | 371 |
| menthol $C_{10}H_{20}O$ [1490-04-6] NOOLISFMXDJSKH-UHFFFAOYSA-N | $3.1\times10^{-1}$ | | Ebert et al. (2023) | ? | 318 |





Table A3.2: Alcohols (ROH) (...continued)

| Substance Formula (Trivial Name) [CAS Registry Number] InChIKey | $H_s^{cp}$ (at $T^\ominus$) $\left[\dfrac{\text{mol}}{\text{m}^3\,\text{Pa}}\right]$ | $\dfrac{\text{d}\ln H_s^{cp}}{\text{d}(1/T)}$ [K] | Reference | Type | Note |
|---|---|---|---|---|---|
| 2-methylisoborneol $C_{11}H_{20}O$ [2371-42-8] LFYXNXGVLGKVCJ-BOBPJJCASA-N | $8.9\times10^{-2}$ 1.6 | 9800 | Ömür-Özbek and Dietrich (2005) Wu et al. (2022a) | M Q | 413 |
| geosmin $C_{12}H_{22}O$ [19700-21-1] JLPUXFOGCDVKGO-GRYCIOLGSA-N | $8.1\times10^{-2}$ 4.1 | 9800 | Ömür-Özbek and Dietrich (2005) Wu et al. (2022a) | M Q | 413 |
| cyclododecanol $C_{12}H_{24}O$ [1724-39-6] SFVWPXMPRCIVOK-UHFFFAOYSA-N | 3.4 6.8 4.5 $3.7\times10^{-1}$ 3.4 8.0 $5.3\times10^{-2}$ 1.6 $2.5\times10^{-1}$ 3.4 | | Altschuh et al. (1999) Keshavarz et al. (2022) Duchowicz et al. (2020) Zhang et al. (2010) Zhang et al. (2010) Zhang et al. (2010) Zhang et al. (2010) Hilal et al. (2008) Modarresi et al. (2007) Duchowicz et al. (2020) | M Q Q Q Q Q Q Q Q ? | 287, 288 287, 289 287, 290 287, 291 67 185, 21 |
| cedrol $C_{15}H_{26}O$ [77-53-2] SVURIXNDRWRAFU-OGMFBOKVSA-N | 6.7 | | Dupeux et al. (2022) | Q | 259 |
| patchoulol $C_{15}H_{26}O$ [5986-55-0] GGHMUJBZYLPWFD-CUZKYEQNSA-N | $1.3\times10^1$ | | Dupeux et al. (2022) | Q | 259 |
| perhydrobisphenol a $C_{15}H_{28}O_2$ [80-04-6] CDBAMNGURPMUTG-UHFFFAOYSA-N | 9.7 $6.1\times10^4$ $3.4\times10^4$ $1.8\times10^2$ | | Zhang et al. (2010) Zhang et al. (2010) Zhang et al. (2010) Zhang et al. (2010) | Q Q Q Q | 287, 288 287, 289 287, 290 287, 291 |
| 3-(5,5,6-trimethyl-2-norbornyl)cyclohexanol $C_{16}H_{28}O$ [3407-42-9] BWVZAZPLUTUBKD-UHFFFAOYSA-N | $6.1\times10^{-1}$ $1.6\times10^1$ $1.1\times10^1$ $2.7\times10^{-1}$ | | Zhang et al. (2010) Zhang et al. (2010) Zhang et al. (2010) Zhang et al. (2010) | Q Q Q Q | 287, 288 287, 289 287, 290 287, 291 |
| 4-(5,5,6-trimethylbicyclo[2.2.1]hept-2-yl)cyclohexan-1-ol $C_{16}H_{28}O$ [66068-84-6] PCFHYANYPQEMPU-UHFFFAOYSA-N | $6.1\times10^{-1}$ $1.8\times10^1$ $4.4\times10^1$ $2.7\times10^{-1}$ | | Zhang et al. (2010) Zhang et al. (2010) Zhang et al. (2010) Zhang et al. (2010) | Q Q Q Q | 287, 288 287, 289 287, 290 287, 291 |





Table A3.2: Alcohols (ROH) (... continued)

| Substance Formula (Trivial Name) [CAS Registry Number] InChIKey | $H_s^{cp}$ (at $T^{\ominus}$) $\left[\dfrac{\text{mol}}{\text{m}^3\,\text{Pa}}\right]$ | $\dfrac{\text{d}\ln H_s^{cp}}{\text{d}(1/T)}$ [K] | Reference | Type | Note |
|---|---|---|---|---|---|
| 4-((1R,2R,4R)-born-2-yl)cyclohexanol | $6.1\times10^{-1}$ | | Zhang et al. (2010) | Q | 287, 288 |
| $C_{16}H_{28}O$ | 9.9 | | Zhang et al. (2010) | Q | 287, 289 |
| [66072-32-0] | $4.3\times10^{1}$ | | Zhang et al. (2010) | Q | 287, 290 |
| LFHQKYSBKVWWOS-UHFFFAOYSA-N | $1.6\times10^{-1}$ | | Zhang et al. (2010) | Q | 287, 291 |
| ethenol | $8.5\times10^{-1}$ | | Wang et al. (2017) | Q | 80, 238 |
| $C_2H_4O$ | 2.0 | | Wang et al. (2017) | Q | 80, 239 |
| (vinyl alcohol) | 1.7 | | Wang et al. (2017) | Q | 80, 240 |
| [557-75-5] | | | | | |
| IMROMDMJAWUWLK-UHFFFAOYSA-N | | | | | |
| 2-propen-1-ol | | 6500 | Plyasunov and Shock (2000) | L | |
| $C_3H_5OH$ | 2.0 | | Lide and Frederikse (1995) | V | |
| (allyl alcohol) | 1.6 | | Yaws (2003) | X | 258 |
| [107-18-6] | 4.3 | 7200 | Goldstein (1982) | X | 298 |
| XXROGKLTLUQVRX-UHFFFAOYSA-N | 2.0 | | Pierotti et al. (1959) | X | 414 |
| | 1.5 | | Dupeux et al. (2022) | Q | 259 |
| | $9.0\times10^{-1}$ | | Keshavarz et al. (2022) | Q | |
| | 4.0 | | Duchowicz et al. (2020) | Q | 299 |
| | $6.9\times10^{-1}$ | | Wang et al. (2017) | Q | 80, 238 |
| | 5.9 | | Wang et al. (2017) | Q | 80, 239 |
| | 2.1 | | Wang et al. (2017) | Q | 80, 240 |
| | 2.0 | | Li et al. (2014) | Q | 241 |
| | 4.9 | | Raventos-Duran et al. (2010) | Q | 242, 243 |
| | 3.9 | | Raventos-Duran et al. (2010) | Q | 244 |
| | 1.6 | | Raventos-Duran et al. (2010) | Q | 245 |
| | 2.8 | | Hilal et al. (2008) | Q | |
| | 3.7 | | Modarresi et al. (2007) | Q | 67 |
| | 2.0 | | Yaffe et al. (2003) | Q | 248, 249 |
| | 3.8 | | Yao et al. (2002) | Q | 229 |
| | 5.1 | | English and Carroll (2001) | Q | 230, 231 |
| | 4.4 | | Katritzky et al. (1998) | Q | |
| | 3.5 | | Nirmalakhandan et al. (1997) | Q | |
| | 4.4 | | Suzuki et al. (1992) | Q | 232 |
| | 3.4 | | Nirmalakhandan and Speece (1988) | Q | |
| | 2.0 | | Duchowicz et al. (2020) | ? | 185, 21 |
| | 1.6 | | Yaws (1999) | ? | 21 |
| | 1.8 | | Yaws and Yang (1992) | ? | 21 |
| | 2.0 | | Abraham et al. (1990) | ? | |
| 2-propyn-1-ol | 3.8 | 7400 | Hiatt (2013) | M | |
| $C_3H_4O$ | 8.6 | | Duchowicz et al. (2020) | V | 186 |
| (propargyl alcohol) | 9.0 | | HSDB (2015) | V | |
| [107-19-7] | 7.0 | | Duchowicz et al. (2020) | Q | |
| TVDSBUOJIPERQY-UHFFFAOYSA-N | 5.4 | | Hilal et al. (2008) | Q | |
| | 7.0 | | Modarresi et al. (2007) | Q | 67 |



Table A3.2: Alcohols (ROH) (. . . continued)

| Substance<br>Formula<br>(Trivial Name)<br>[CAS Registry Number]<br>InChIKey | $H_s^{cp}$<br>(at $T^{\ominus}$)<br>$\left[\dfrac{\text{mol}}{\text{m}^3\,\text{Pa}}\right]$ | $\dfrac{\text{d}\ln H_s^{cp}}{\text{d}(1/T)}$<br><br>[K] | Reference | Type | Note |
|---|---|---|---|---|---|
| 2-buten-1-ol<br>$CH_3CHCHCH_2OH$<br>[6117-91-5]<br>WCASXYBKJHWFMY-UHFFFAOYSA-N | <br>3.1<br>3.9<br>1.2<br>2.7<br>3.0 | 6900 | Plyasunov and Shock (2000)<br>Raventos-Duran et al. (2010)<br>Raventos-Duran et al. (2010)<br>Raventos-Duran et al. (2010)<br>Hilal et al. (2008)<br>Saxena and Hildemann (1996) | L<br>Q<br>Q<br>Q<br>Q<br>E | <br>242, 243<br>244<br>245<br><br>401 |
| 3-buten-1-ol<br>$C_4H_8O$<br>[627-27-0]<br>ZSPTYLOMNJNZNG-UHFFFAOYSA-N | | 7000 | Plyasunov and Shock (2000) | L | |
| 4-penten-1-ol<br>$C_5H_{10}O$<br>[821-09-0]<br>LQAVWYMTUMSFBE-UHFFFAOYSA-N | | 7300 | Plyasunov and Shock (2000) | L | |
| 3-pentyn-1-ol<br>$C_5H_8O$<br>[10229-10-4]<br>IDYNOORNKYEHHO-UHFFFAOYSA-N | | 7800 | Plyasunov and Shock (2000) | L | |
| 2-methyl-3-buten-2-ol<br>$C_5H_9O$<br>[115-18-4]<br>HNVRRHSXBLFLIG-UHFFFAOYSA-N | $6.4\times10^{-1}$<br>$4.7\times10^{-1}$<br>1.6<br>$7.9\times10^{-1}$<br>$3.6\times10^{-1}$<br>1.2<br>$7.3\times10^{-1}$<br>2.5<br>$6.2\times10^{-1}$<br>$9.9\times10^{-1}$<br>$6.0\times10^{-1}$<br>1.6<br>$4.7\times10^{-1}$ | | Iraci et al. (1999)<br>Altschuh et al. (1999)<br>Keshavarz et al. (2022)<br>Duchowicz et al. (2020)<br>Wang et al. (2017)<br>Wang et al. (2017)<br>Wang et al. (2017)<br>Raventos-Duran et al. (2010)<br>Raventos-Duran et al. (2010)<br>Raventos-Duran et al. (2010)<br>Hilal et al. (2008)<br>Modarresi et al. (2007)<br>Duchowicz et al. (2020) | M<br>M<br>Q<br>Q<br>Q<br>Q<br>Q<br>Q<br>Q<br>Q<br>Q<br>Q<br>? | 38<br><br><br>299<br>80, 238<br>80, 239<br>80, 240<br>242, 243<br>244<br>245<br><br>67<br>185, 21 |
| 2-methyl-3-butyn-2-ol<br>$C_5H_8O$<br>[115-19-5]<br>CEBKHWWANWSNTI-UHFFFAOYSA-N | 2.5<br>1.6<br>1.5<br>1.0<br>8.3<br>2.5 | | Altschuh et al. (1999)<br>Keshavarz et al. (2022)<br>Duchowicz et al. (2020)<br>Hilal et al. (2008)<br>Modarresi et al. (2007)<br>Duchowicz et al. (2020) | M<br>Q<br>Q<br>Q<br>Q<br>? | <br><br><br><br>67<br>185, 21 |
| 3-methyl-1-pentyn-3-ol<br>$C_6H_{10}O$<br>(meparfynol; methyl pentynol)<br>[77-75-8]<br>QXLPXWSKPNOQLE-UHFFFAOYSA-N | $9.9\times10^{-1}$<br>7.0<br>4.3 | | Hilal et al. (2008)<br>Modarresi et al. (2007)<br>Katritzky et al. (1998) | Q<br>Q<br>Q | <br>67<br> |



Table A3.2: Alcohols (ROH) (...continued)

| Substance Formula (Trivial Name) [CAS Registry Number] InChIKey | $H_s^{cp}$ (at $T^\ominus$) $\left[\dfrac{\text{mol}}{\text{m}^3\,\text{Pa}}\right]$ | $\dfrac{\text{d}\ln H_s^{cp}}{\text{d}(1/T)}$ [K] | Reference | Type | Note |
|---|---|---|---|---|---|
| 2-cyclohexen-1-ol $C_6H_{10}O$ [822-67-3] PQANGXXSEABURG-UHFFFAOYSA-N | | 8500 | Plyasunov and Shock (2000) | L | |
| bicyclo[2.2.1]heptan-2-ol $C_7H_{12}O$ (norborneol) [1632-68-4] ZQTYQMYDIHMKQB-UHFFFAOYSA-N | 2.2 | 5000 | van Roon et al. (2005) | V | |
| 2-octen-1-ol $C_8H_{16}O$ [22104-78-5] AYQPVPFZWIQERS-UHFFFAOYSA-N | $7.7\times10^{-1}$ | | Wu et al. (2022a) | Q | 413 |
| 1-octen-3-ol $C_8H_{16}O$ [3391-86-4] VSMOENVRRABVKN-UHFFFAOYSA-N | $1.9\times10^{-1}$ $2.5\times10^{-1}$ $1.3\times10^{-1}$ | 7900 | Wu et al. (2022a) Druaux et al. (1998) Roberts and Pollien (1997) | M M M | |
| 3,7-dimethyl-6-octen-1-ol $C_{10}H_{20}O$ (citronellol) [106-22-9] QMVPMAAFGQKVCJ-UHFFFAOYSA-N | 5.2 $4.9\times10^{-1}$ | | Martins et al. (2017) Dupeux et al. (2022) | V Q | 315 259 |
| 3,7-dimethyl-1,6-octadien-3-ol $C_{10}H_{18}O$ (linalool) [78-70-6] CDOSHBSSFJOMGT-UHFFFAOYSA-N | $2.0\times10^{-1}$ $3.8\times10^{-1}$ $4.6\times10^{-1}$ $4.0\times10^{-1}$ $4.8\times10^{-1}$ $4.8\times10^{-1}$ $2.1\times10^{-1}$ 1.6 $1.3\times10^{-1}$ $3.3\times10^{-1}$ $2.5\times10^{-1}$ 2.5 $6.2\times10^{-1}$ $2.5\times10^{-1}$ $6.9\times10^{-1}$ 1.4 $1.5\times10^{-2}$ $4.6\times10^{-1}$ | 4400 14000 | Leng et al. (2013) Copolovici and Niinemets (2007) Altschuh et al. (1999) Martins et al. (2017) Copolovici and Niinemets (2005) Niinemets and Reichstein (2002) Li et al. (1998) Dupeux et al. (2022) Keshavarz et al. (2022) Duchowicz et al. (2020) Savary et al. (2014) Raventos-Duran et al. (2010) Raventos-Duran et al. (2010) Raventos-Duran et al. (2010) Hilal et al. (2008) Modarresi et al. (2007) Hertel and Sommer (2006) Duchowicz et al. (2020) | M M M V V V V Q Q Q Q Q Q Q Q Q Q ? | 315 259 184 271, 243 244 245 67 415 185, 21 |
| linalool oxide $C_{10}H_{18}O_2$ [1365-19-1] BXOURKNXQXLKRK-UHFFFAOYSA-N | $3.0\times10^2$ | | Dupeux et al. (2022) | Q | 259 |



Table A3.2: Alcohols (ROH) (. . . continued)

| Substance Formula (Trivial Name) [CAS Registry Number] InChIKey | $H_s^{cp}$ (at $T^{\ominus}$) $\left[\dfrac{\text{mol}}{\text{m}^3\,\text{Pa}}\right]$ | $\dfrac{\text{d}\ln H_s^{cp}}{\text{d}(1/T)}$ [K] | Reference | Type | Note |
|---|---|---|---|---|---|
| (E)-3,7-dimethyl-2,6-octadien-1-ol $C_{10}H_{18}O$ (geraniol) [106-24-1] GLZPCOQZEFWAFX-JXMROGBWSA-N | 3.1 1.6 $1.7\times10^{-1}$ | | Martins et al. (2017) Dupeux et al. (2022) HSDB (2015) | V Q Q | 315 259 99 |
| (Z)-3,7-dimethyl-2,6-octadien-1-ol $C_{10}H_{18}O$ [106-25-2] GLZPCOQZEFWAFX-YFHOEESVSA-N | $8.6\times10^{-1}$ $2.1\times10^{-1}$ | | Duchowicz et al. (2020) Duchowicz et al. (2020) | V Q | 186 |
| tricyclo[3.3.1.1(3,7)]decan-1-ol $C_{10}H_{16}O$ (1-adamantanol) [768-95-6] VLLNJDMHDJRNFK-UHFFFAOYSA-N | 6.0 | 5300 | van Roon et al. (2005) | V | |
| 3,7,11-trimethyl-2,6,10-dodecatrien-1-ol $C_{15}H_{26}O$ (farnesol) [4602-84-0] CRDAMVZIKSXKFV-YFVJMOTDSA-N | $3.9\times10^{-2}$ | | HSDB (2015) | Q | 99 |
| (Z)-9-octadecen-1-ol $C_{18}H_{36}O$ (oleyl alcohol) [143-28-2] ALSTYHKOOCGGFT-KTKRTIGZSA-N | $2.1\times10^{-2}$ | | HSDB (2015) | Q | 99 |
| (3E,13Z)-octadeca-3,13-dien-1-ol $C_{18}H_{34}O$ [66410-28-4] QBNCGBJHGBGHLS-IYUNJCAYSA-N | $2.4\times10^{-1}$ | | Ebert et al. (2023) | ? | 318 |
| dihydroabietyl alcohol $C_{20}H_{34}O$ [26266-77-3] FLMIYUXOBAUKJM-UHFFFAOYSA-N | $1.9\times10^{-1}$ $2.4\times10^{1}$ $2.6\times10^{1}$ $2.0\times10^{-1}$ | | Zhang et al. (2010) Zhang et al. (2010) Zhang et al. (2010) Zhang et al. (2010) | Q Q Q Q | 287, 288 287, 289 287, 290 287, 291 |
| ethylestrenol $C_{20}H_{32}O$ [965-90-2] AOXRBFRFYPMWLR-UHFFFAOYSA-N | $4.3\times10^{-1}$ | | HSDB (2015) | Q | 99 |
| trans-phytol $C_{20}H_{40}O$ [150-86-7] BOTWFXYSPFMFNR-PYDDKJGSSA-N | $1.0\times10^{-2}$ | | Ebert et al. (2023) | ? | 365 |



Table A3.2: Alcohols (ROH) (...continued)

| Substance<br>Formula<br>(Trivial Name)<br>[CAS Registry Number]<br>InChIKey | $H_s^{cp}$<br>(at $T^\ominus$)<br>$\left[\dfrac{\mathrm{mol}}{\mathrm{m^3\,Pa}}\right]$ | $\dfrac{\mathrm{d}\ln H_s^{cp}}{\mathrm{d}(1/T)}$<br><br>[K] | Reference | Type | Note |
|---|---|---|---|---|---|
| hydroxybenzene | $1.8\times10^1$ | 3000 | Schwardt et al. (2021) | L | 1 |
| $C_6H_5OH$ | $2.2\times10^1$ | 8900 | Brockbank (2013) | L | 1 |
| (phenol) | $2.2\times10^1$ | 9800 | Ji et al. (2008) | M | |
| [108-95-2] | $2.8\times10^1$ | 2700 | Guo and Brimblecombe (2007) | M | |
| ISWSIDIOOBJBQZ-UHFFFAOYSA-N | 6.4 | 7700 | Feigenbrugel et al. (2004b) | M | |
| | $3.0\times10^1$ | 5900 | Harrison et al. (2002) | M | |
| | $1.9\times10^1$ | | Sheikheldin et al. (2001) | M | 12 |
| | $>4.2$ | | Altschuh et al. (1999) | M | |
| | $8.1\times10^1$ | 7400 | Tabai et al. (1997) | M | 11 |
| | 4.2 | | Heal et al. (1995) | M | 373 |
| | $1.6\times10^1$ | 6000 | Dohnal and Fenclová (1995) | M | |
| | $1.5\times10^1$ | | Tremp et al. (1993) | M | 12 |
| | $1.7\times10^1$ | 6100 | Abd-El-Bary et al. (1986) | M | |
| | 7.6 | | Warner et al. (1980) | M | |
| | $2.0\times10^1$ | | Mackay et al. (2006c) | V | |
| | $2.5\times10^1$ | | Lide and Frederikse (1995) | V | |
| | $1.9\times10^1$ | | Mackay et al. (1995) | V | |
| | $1.9\times10^1$ | | Shiu et al. (1994) | V | |
| | 3.4 | | Hwang et al. (1992) | V | |
| | $1.1\times10^1$ | | Riederer (1990) | V | |
| | $9.0\times10^1$ | | Leuenberger et al. (1985) | V | 416 |
| | 4.8 | | Hine and Weimar (1965) | R | |
| | $2.8\times10^1$ | 6800 | Parsons et al. (1971) | T | 417 |
| | $1.3\times10^1$ | | Yaws (2003) | X | 258 |
| | 1.9 | 3600 | Janini and Quaddora (1986) | X | 298 |
| | $1.9\times10^1$ | 7300 | Goldstein (1982) | X | 298 |
| | $2.5\times10^1$ | | Howard (1989) | X | 418 |
| | $3.0\times10^1$ | | Gaffney and Senum (1984) | X | 389 |
| | $3.7\times10^1$ | | McCarty (1980) | X | 368 |
| | $2.5\times10^1$ | | Schüürmann (2000) | C | 21 |
| | 7.6 | | Shiu et al. (1994) | C | |
| | 7.6 | | Smith et al. (1993) | C | |
| | $2.1\times10^1$ | | Ryan et al. (1988) | C | |
| | 7.6 | | Shen (1982) | C | |
| | $1.8\times10^1$ | | Dupeux et al. (2022) | Q | 259 |
| | $6.4\times10^1$ | | Keshavarz et al. (2022) | Q | |
| | $4.2\times10^1$ | | Duchowicz et al. (2020) | Q | 184 |
| | $6.3\times10^1$ | | Wang et al. (2017) | Q | 80, 238 |
| | $1.1\times10^1$ | | Wang et al. (2017) | Q | 80, 239 |
| | $3.6\times10^1$ | | Wang et al. (2017) | Q | 80, 240 |
| | $2.5\times10^1$ | | Li et al. (2014) | Q | 241 |
| | 9.9 | | Raventos-Duran et al. (2010) | Q | 242, 243 |
| | 7.8 | | Raventos-Duran et al. (2010) | Q | 244 |
| | $1.6\times10^1$ | | Raventos-Duran et al. (2010) | Q | 245 |
| | 4.4 | | Hilal et al. (2008) | Q | |
| | $1.8\times10^1$ | | Modarresi et al. (2007) | Q | 67 |
| | | 6200 | Kühne et al. (2005) | Q | |



Table A3.2: Alcohols (ROH) (...continued)

| Substance Formula (Trivial Name) [CAS Registry Number] InChIKey | $H_s^{cp}$ (at $T^\ominus$) $\left[\dfrac{\text{mol}}{\text{m}^3\,\text{Pa}}\right]$ | $\dfrac{\text{d}\ln H_s^{cp}}{\text{d}(1/T)}$ [K] | Reference | Type | Note |
|---|---|---|---|---|---|
| | $3.0\times10^1$ | | Yaffe et al. (2003) | Q | 248, 249 |
| | $2.9\times10^1$ | | English and Carroll (2001) | Q | 230, 231 |
| | 6.9 | | Katritzky et al. (1998) | Q | |
| | $2.0\times10^1$ | | Russell et al. (1992) | Q | 279 |
| | $2.0\times10^1$ | | Suzuki et al. (1992) | Q | 232 |
| | 9.9 | | Nirmalakhandan and Speece (1988) | Q | |
| | $3.0\times10^1$ | | Duchowicz et al. (2020) | ? | 185, 21 |
| | | 5400 | Kühne et al. (2005) | ? | |
| | $1.3\times10^1$ | | Yaws (1999) | ? | 21 |
| | $1.6\times10^1$ | | Abraham et al. (1990) | ? | |
| (hydroxymethyl)-benzene $C_6H_5CH_2OH$ (benzyl alcohol) [100-51-6] WVDDGKGOMKODPV-UHFFFAOYSA-N | $1.7\times10^1$ | | Chao et al. (2017) | M | |
| | $>3.7\times10^1$ | | Altschuh et al. (1999) | M | |
| | $2.5\times10^1$ | | Chao et al. (2017) | V | |
| | $6.2\times10^{-2}$ | | Mackay et al. (2006c) | V | |
| | $6.2\times10^{-2}$ | | Mackay et al. (1995) | V | |
| | $2.9\times10^1$ | | Abraham et al. (1994a) | R | |
| | $3.8\times10^1$ | | Yaws (2003) | X | 258 |
| | $3.8\times10^1$ | | Yaws (2003) | X | 237 |
| | $2.5\times10^1$ | | Howard (1993) | X | 412 |
| | $1.6\times10^1$ | | Dupeux et al. (2022) | Q | 259 |
| | $2.3\times10^1$ | | Keshavarz et al. (2022) | Q | |
| | $2.3\times10^1$ | | Duchowicz et al. (2020) | Q | 299 |
| | $3.8\times10^1$ | | Wang et al. (2017) | Q | 80, 238 |
| | $4.9\times10^1$ | | Wang et al. (2017) | Q | 80, 239 |
| | $3.8\times10^1$ | | Wang et al. (2017) | Q | 80, 240 |
| | $8.6\times10^1$ | | Gharagheizi et al. (2012) | Q | |
| | $9.9\times10^1$ | | Raventos-Duran et al. (2010) | Q | 242, 243 |
| | $3.9\times10^1$ | | Raventos-Duran et al. (2010) | Q | 244 |
| | $4.9\times10^1$ | | Raventos-Duran et al. (2010) | Q | 245 |
| | $4.1\times10^1$ | | Gharagheizi et al. (2010) | Q | 246 |
| | $2.2\times10^1$ | | Hilal et al. (2008) | Q | |
| | $5.5\times10^1$ | | Modarresi et al. (2007) | Q | 67 |
| | $1.1\times10^1$ | | Yao et al. (2002) | Q | 229 |
| | $2.1\times10^1$ | | English and Carroll (2001) | Q | 230, 231 |
| | $6.9\times10^1$ | | Nirmalakhandan et al. (1997) | Q | |
| | $8.9\times10^1$ | | Saxena and Hildemann (1996) | E | 401 |
| | $2.9\times10^1$ | | Duchowicz et al. (2020) | ? | 185, 21 |
| | $2.9\times10^1$ | | HSDB (2015) | ? | 419 |
| | $3.8\times10^1$ | | Yaws (1999) | ? | 21 |
| | $1.8\times10^1$ | | Abraham et al. (1990) | ? | |



Table A3.2: Alcohols (ROH) (...continued)

| Substance Formula (Trivial Name) [CAS Registry Number] InChIKey | $H_s^{cp}$ (at $T^{\ominus}$) $\left[\dfrac{\text{mol}}{\text{m}^3\,\text{Pa}}\right]$ | $\dfrac{\text{d}\ln H_s^{cp}}{\text{d}(1/T)}$ [K] | Reference | Type | Note |
|---|---|---|---|---|---|
| 1-hydroxy-2-methylbenzene | 6.7 | 7400 | Brockbank (2013) | L | 1 |
| $HOC_6H_4CH_3$ | 6.5 | | Chao et al. (2017) | M | |
| (2-cresol; $o$-cresol) | 4.2 | 8500 | Feigenbrugel et al. (2004b) | M | |
| [95-48-7] | $1.1\times10^1$ | 6700 | Harrison et al. (2002) | M | |
| QWVGKYWNOKOFNN-UHFFFAOYSA-N | 6.3 | | Altschuh et al. (1999) | M | |
| | 5.6 | 5800 | Dohnal and Fenclová (1995) | M | |
| | 7.1 | | Tremp et al. (1993) | M | 12 |
| | 8.2 | 7300 | Parsons et al. (1972) | M | 417 |
| | 5.8 | | Chao et al. (2017) | V | |
| | | | Mackay et al. (2006c) | V | 420 |
| | 6.2 | | Lide and Frederikse (1995) | V | |
| | 6.4 | | Mackay et al. (1995) | V | |
| | $3.5\times10^1$ | | Leuenberger et al. (1985) | V | 416 |
| | 8.8 | | Yaws (2003) | X | 258 |
| | 2.6 | 4600 | Janini and Quaddora (1986) | X | 298 |
| | 6.2 | | Howard (1989) | X | 418 |
| | 8.2 | | Gaffney and Senum (1984) | X | 389 |
| | 8.3 | | Schüürmann (2000) | C | 21 |
| | 8.4 | | Dupeux et al. (2022) | Q | 259 |
| | 5.9 | | Keshavarz et al. (2022) | Q | |
| | $2.3\times10^1$ | | Duchowicz et al. (2020) | Q | |
| | $3.9\times10^1$ | | Wang et al. (2017) | Q | 80, 238 |
| | $1.2\times10^1$ | | Wang et al. (2017) | Q | 80, 239 |
| | $1.1\times10^1$ | | Wang et al. (2017) | Q | 80, 240 |
| | 8.0 | | Li et al. (2014) | Q | 241 |
| | $1.2\times10^1$ | | Gharagheizi et al. (2012) | Q | |
| | 5.3 | | Hilal et al. (2008) | Q | |
| | 6.2 | | Modarresi et al. (2007) | Q | 67 |
| | | 6500 | Kühne et al. (2005) | Q | |
| | 8.8 | | Yaffe et al. (2003) | Q | 248, 249 |
| | 9.9 | | Yao et al. (2002) | Q | 229, 267 |
| | 9.5 | | English and Carroll (2001) | Q | 230, 231 |
| | 4.6 | | Katritzky et al. (1998) | Q | |
| | $1.5\times10^1$ | | Suzuki et al. (1992) | Q | 232 |
| | 7.2 | | Nirmalakhandan and Speece (1988) | Q | |
| | 8.2 | | Duchowicz et al. (2020) | ? | 185, 21 |
| | | 8100 | Kühne et al. (2005) | ? | |
| | 5.8 | | Yaws (1999) | ? | 21, 12 |
| | $1.2\times10^1$ | | Yaws and Yang (1992) | ? | 21, 12 |
| | 8.0 | | Abraham et al. (1990) | ? | |
| 1-hydroxy-3-methylbenzene | $1.2\times10^1$ | 6200 | Brockbank (2013) | L | 1 |
| $HOC_6H_4CH_3$ | 7.9 | 9000 | Feigenbrugel et al. (2004b) | M | |
| (3-cresol; $m$-cresol) | $1.2\times10^1$ | | Altschuh et al. (1999) | M | |
| [108-39-4] | $1.2\times10^1$ | 6000 | Dohnal and Fenclová (1995) | M | |
| RLSSMJSEOOYNOY-UHFFFAOYSA-N | $1.3\times10^1$ | | Mackay et al. (2006c) | V | |
| | $1.2\times10^1$ | | Schüürmann (2000) | V | |
| | $1.1\times10^1$ | | Lide and Frederikse (1995) | V | |



Table A3.2: Alcohols (ROH) (. . . continued)

| Substance Formula (Trivial Name) [CAS Registry Number] InChIKey | $H_s^{cp}$ (at $T^{\ominus}$) $\left[\dfrac{\text{mol}}{\text{m}^3\,\text{Pa}}\right]$ | $\dfrac{\text{d}\ln H_s^{cp}}{\text{d}(1/T)}$ [K] | Reference | Type | Note |
|---|---|---|---|---|---|
|  | $1.1\times10^1$ |  | Mackay et al. (1995) | V |  |
|  | $1.1\times10^1$ |  | Meylan and Howard (1991) | V |  |
|  | $4.9\times10^1$ |  | Leuenberger et al. (1985) | V | 416 |
|  | $2.2\times10^1$ |  | Yaws (2003) | X | 258 |
|  | 6.1 | 7700 | Janini and Quaddora (1986) | X | 298 |
|  | $1.1\times10^1$ |  | Howard (1989) | X | 418 |
|  | $1.8\times10^1$ |  | Dupeux et al. (2022) | Q | 259 |
|  | 5.9 |  | Keshavarz et al. (2022) | Q |  |
|  | $2.3\times10^1$ |  | Duchowicz et al. (2020) | Q | 184 |
|  | $1.2\times10^1$ |  | Gharagheizi et al. (2012) | Q |  |
|  | 3.9 |  | Hilal et al. (2008) | Q |  |
|  | $1.1\times10^1$ |  | Modarresi et al. (2007) | Q | 67 |
|  |  | 6500 | Kühne et al. (2005) | Q |  |
|  | $1.2\times10^1$ |  | Yaffe et al. (2003) | Q | 248, 249 |
|  | $1.0\times10^1$ |  | Yao et al. (2002) | Q | 229 |
|  | $1.7\times10^1$ |  | Katritzky et al. (1998) | Q |  |
|  | $1.6\times10^1$ |  | Meylan and Howard (1991) | Q |  |
|  | $1.2\times10^1$ |  | Duchowicz et al. (2020) | ? | 185, 21 |
|  |  | 6500 | Kühne et al. (2005) | ? |  |
|  | $1.4\times10^1$ |  | Yaws (1999) | ? | 21, 12 |
|  | $1.4\times10^1$ |  | Yaws and Yang (1992) | ? | 21, 12 |
|  | 4.3 |  | Abraham et al. (1990) | ? |  |
| 1-hydroxy-4-methylbenzene $HOC_6H_4CH_3$ (4-cresol; $p$-cresol) [106-44-5] IWDCLRJOBJJRNH-UHFFFAOYSA-N | $1.3\times10^1$ | 6600 | Brockbank (2013) | L | 1 |
|  | $1.0\times10^1$ | 9300 | Feigenbrugel et al. (2004b) | M |  |
|  | >2.9 |  | Altschuh et al. (1999) | M |  |
|  | $1.3\times10^1$ | 6100 | Dohnal and Fenclová (1995) | M |  |
|  | $1.3\times10^1$ |  | Tremp et al. (1993) | M | 12 |
|  | $1.3\times10^1$ | 7200 | Parsons et al. (1972) | M | 417 |
|  | $1.8\times10^1$ |  | Mackay et al. (2006c) | V |  |
|  | $1.0\times10^1$ |  | Lide and Frederikse (1995) | V |  |
|  | $1.5\times10^1$ |  | Mackay et al. (1995) | V |  |
|  | $4.5\times10^1$ |  | Leuenberger et al. (1985) | V | 416 |
|  | 1.2 |  | Smith and Bomberger (1980) | V | 24 |
|  | $2.2\times10^1$ |  | Yaws (2003) | X | 258 |
|  | 5.2 | 4600 | Janini and Quaddora (1986) | X | 298 |
|  | $1.0\times10^1$ |  | Howard (1989) | X | 418 |
|  | 9.9 |  | Gaffney and Senum (1984) | X | 389 |
|  | $1.3\times10^1$ |  | Schüürmann (2000) | C | 21 |
|  | $1.6\times10^1$ |  | Dupeux et al. (2022) | Q | 259 |
|  | 5.9 |  | Keshavarz et al. (2022) | Q |  |
|  | $2.3\times10^1$ |  | Duchowicz et al. (2020) | Q | 184 |
|  | $1.8\times10^1$ |  | Gharagheizi et al. (2012) | Q |  |
|  | 4.2 |  | Hilal et al. (2008) | Q |  |
|  | $1.7\times10^1$ |  | Modarresi et al. (2007) | Q | 67 |
|  |  | 6500 | Kühne et al. (2005) | Q |  |
|  | 8.8 |  | Yaffe et al. (2003) | Q | 248, 272 |
|  | $1.5\times10^1$ |  | Yao et al. (2002) | Q | 229 |



Table A3.2: Alcohols (ROH) (...continued)

| Substance Formula (Trivial Name) [CAS Registry Number] InChIKey | $H_s^{cp}$ (at $T^\ominus$) $\left[\dfrac{\text{mol}}{\text{m}^3\,\text{Pa}}\right]$ | $\dfrac{\text{d}\ln H_s^{cp}}{\text{d}(1/T)}$ [K] | Reference | Type | Note |
|---|---|---|---|---|---|
| | $1.7\times10^1$ | | English and Carroll (2001) | Q | 230, 231 |
| | $1.5\times10^1$ | | Katritzky et al. (1998) | Q | |
| | $1.4\times10^1$ | | Suzuki et al. (1992) | Q | 232 |
| | 7.0 | | Nirmalakhandan and Speece (1988) | Q | |
| | 9.9 | | Duchowicz et al. (2020) | ? | 185, 21 |
| | | 6000 | Kühne et al. (2005) | ? | |
| | $1.3\times10^1$ | | Yaws (1999) | ? | 21, 12 |
| | $2.5\times10^1$ | | Yaws and Yang (1992) | ? | 21, 12 |
| | $1.3\times10^1$ | | Abraham et al. (1990) | ? | |
| 1-hydroxy-2,3-dimethylbenzene $C_8H_{10}O$ (2,3-xylenol; 2,3-dimethylphenol) [526-75-0] QWBBPBRQALCEIZ-UHFFFAOYSA-N | 6.7 | 6100 | Brockbank (2013) | L | 1 |
| | 9.3 | | Sheikheldin et al. (2001) | M | 12 |
| | $1.0\times10^1$ | 6800 | Dohnal and Fenclová (1995) | M | |
| | 3.2 | | HSDB (2015) | V | |
| | $1.8\times10^1$ | | Mackay et al. (2006c) | V | |
| | $1.9\times10^1$ | | Mackay et al. (1995) | V | |
| | $4.9\times10^1$ | | Leuenberger et al. (1985) | V | 416 |
| | $1.3\times10^1$ | | Abraham et al. (1994a) | R | |
| | 7.9 | | Keshavarz et al. (2022) | Q | |
| | $1.2\times10^1$ | | Duchowicz et al. (2020) | Q | 184 |
| | $2.2\times10^1$ | | Wang et al. (2017) | Q | 80, 238 |
| | $1.4\times10^1$ | | Wang et al. (2017) | Q | 80, 239 |
| | 9.6 | | Wang et al. (2017) | Q | 80, 240 |
| | 6.2 | | Raventos-Duran et al. (2010) | Q | 242, 243 |
| | 7.8 | | Raventos-Duran et al. (2010) | Q | 244 |
| | $1.6\times10^1$ | | Raventos-Duran et al. (2010) | Q | 245 |
| | 5.8 | | Hilal et al. (2008) | Q | |
| | 5.2 | | Modarresi et al. (2007) | Q | 67 |
| | $1.4\times10^1$ | | Yaffe et al. (2003) | Q | 248, 249 |
| | $1.2\times10^1$ | | English and Carroll (2001) | Q | 230, 231 |
| | $4.6\times10^1$ | | Nirmalakhandan et al. (1997) | Q | |
| | $1.3\times10^1$ | | Duchowicz et al. (2020) | ? | 185, 21 |
| 1-hydroxy-2,4-dimethylbenzene $C_8H_{10}O$ (2,4-xylenol; 2,4-dimethylphenol) [105-67-9] KUFFULVDNCHOFZ-UHFFFAOYSA-N | 3.5 | 5500 | Brockbank (2013) | L | 1 |
| | 6.6 | | Sheikheldin et al. (2001) | M | 12 |
| | 4.9 | 6100 | Dohnal and Fenclová (1995) | M | |
| | $1.9\times10^{-3}$ | -3200 | Ashworth et al. (1988) | M | 33, 278 |
| | 5.5 | | Mackay et al. (2006c) | V | |
| | $1.6\times10^1$ | | Lide and Frederikse (1995) | V | |
| | 5.5 | | Mackay et al. (1995) | V | |
| | $5.5\times10^{-1}$ | | Hwang et al. (1992) | V | |
| | 4.9 | | Meylan and Howard (1991) | V | |
| | $1.6\times10^1$ | | Leuenberger et al. (1985) | V | 416 |
| | $1.0\times10^1$ | | Abraham et al. (1994a) | R | |
| | 4.7 | | Yaws (2003) | X | 258 |
| | 4.1 | 6600 | Goldstein (1982) | X | 298 |
| | $1.6\times10^1$ | | Howard (1989) | X | 418 |
| | $5.8\times10^{-1}$ | | Smith et al. (1993) | C | |





Table A3.2: Alcohols (ROH) (...continued)

| Substance Formula (Trivial Name) [CAS Registry Number] InChIKey | $H_s^{cp}$ (at $T^\ominus$) $\left[\dfrac{\text{mol}}{\text{m}^3\,\text{Pa}}\right]$ | $\dfrac{\text{d}\ln H_s^{cp}}{\text{d}(1/T)}$ [K] | Reference | Type | Note |
|---|---|---|---|---|---|
| | $5.4\times10^{-1}$ | | Ryan et al. (1988) | C | |
| | $1.7\times10^{1}$ | | Petrasek et al. (1983) | C | |
| | 6.7 | | Dupeux et al. (2022) | Q | 259 |
| | 7.9 | | Keshavarz et al. (2022) | Q | |
| | $1.2\times10^{1}$ | | Duchowicz et al. (2020) | Q | 299 |
| | $2.2\times10^{1}$ | | Wang et al. (2017) | Q | 80, 238 |
| | $1.2\times10^{1}$ | | Wang et al. (2017) | Q | 80, 239 |
| | 8.7 | | Wang et al. (2017) | Q | 80, 240 |
| | 4.6 | | Gharagheizi et al. (2012) | Q | |
| | 6.2 | | Raventos-Duran et al. (2010) | Q | 242, 243 |
| | 6.2 | | Raventos-Duran et al. (2010) | Q | 244 |
| | $1.6\times10^{1}$ | | Raventos-Duran et al. (2010) | Q | 245 |
| | 5.1 | | Hilal et al. (2008) | Q | |
| | 4.3 | | Modarresi et al. (2007) | Q | 67 |
| | $1.1\times10^{1}$ | | Yaffe et al. (2003) | Q | 248, 249 |
| | $1.6\times10^{1}$ | | English and Carroll (2001) | Q | 230, 231 |
| | 6.7 | | Katritzky et al. (1998) | Q | |
| | $4.6\times10^{1}$ | | Nirmalakhandan et al. (1997) | Q | |
| | $1.4\times10^{1}$ | | Meylan and Howard (1991) | Q | |
| | $1.0\times10^{1}$ | | Duchowicz et al. (2020) | ? | 185, 21 |
| 1-hydroxy-2,5-dimethylbenzene $C_8H_{10}O$ (2,5-xylenol; 2,5-dimethylphenol) [95-87-4] NKTOLZVEWDHZMU-UHFFFAOYSA-N | 8.2 | 8700 | Brockbank (2013) | L | 1 |
| | 7.5 | 6800 | Dohnal and Fenclová (1995) | M | |
| | 1.4 | | HSDB (2015) | V | |
| | 7.5 | | Mackay et al. (2006c) | V | |
| | 7.4 | | Mackay et al. (1995) | V | |
| | $3.8\times10^{1}$ | | Leuenberger et al. (1985) | V | 416 |
| | 8.8 | | Abraham et al. (1994a) | R | |
| | 1.9 | | Yaws (2003) | X | 258 |
| | 5.7 | | Dupeux et al. (2022) | Q | 259 |
| | 7.9 | | Keshavarz et al. (2022) | Q | |
| | $1.2\times10^{1}$ | | Duchowicz et al. (2020) | Q | 299 |
| | $2.2\times10^{1}$ | | Wang et al. (2017) | Q | 80, 238 |
| | $1.2\times10^{1}$ | | Wang et al. (2017) | Q | 80, 239 |
| | 8.9 | | Wang et al. (2017) | Q | 80, 240 |
| | 6.2 | | Raventos-Duran et al. (2010) | Q | 242, 243 |
| | 6.2 | | Raventos-Duran et al. (2010) | Q | 244 |
| | $1.6\times10^{1}$ | | Raventos-Duran et al. (2010) | Q | 245 |
| | 5.2 | | Hilal et al. (2008) | Q | |
| | 4.8 | | Modarresi et al. (2007) | Q | 67 |
| | 3.1 | | Yaffe et al. (2003) | Q | 248, 272 |
| | $1.4\times10^{1}$ | | English and Carroll (2001) | Q | 230, 274 |
| | $4.6\times10^{1}$ | | Nirmalakhandan et al. (1997) | Q | |
| | 8.8 | | Duchowicz et al. (2020) | ? | 185, 21 |



Table A3.2: Alcohols (ROH) (...continued)

| Substance<br>Formula<br>(Trivial Name)<br>[CAS Registry Number]<br>InChIKey | $H_s^{cp}$<br>(at $T^\ominus$)<br>$\left[\dfrac{\text{mol}}{\text{m}^3\,\text{Pa}}\right]$ | $\dfrac{\text{d}\ln H_s^{cp}}{\text{d}(1/T)}$<br><br>[K] | Reference | Type | Note |
|---|---|---|---|---|---|
| 1-hydroxy-2,6-dimethylbenzene | 1.6 | 6300 | Brockbank (2013) | L | 1 |
| $C_8H_{10}O$ | 2.3 | 6200 | Dohnal and Fenclová (1995) | M | |
| (2,6-xylenol; 2,6-dimethylphenol) | 1.3 | | Hawthorne et al. (1985) | M | |
| [576-26-1] | 2.5 | | Mackay et al. (2006c) | V | |
| NXXYKOUNUYWIHA-UHFFFAOYSA-N | 2.6 | | Mackay et al. (1995) | V | |
| | 2.6 | | Shiu et al. (1994) | V | |
| | 5.2 | | Leuenberger et al. (1985) | V | 416 |
| | 2.9 | | Abraham et al. (1994a) | R | |
| | 1.4 | | Yaws (2003) | X | 258 |
| | 1.6 | | Dupeux et al. (2022) | Q | 259 |
| | 7.9 | | Keshavarz et al. (2022) | Q | |
| | 3.0 | | Duchowicz et al. (2020) | Q | 299 |
| | 6.2 | | Raventos-Duran et al. (2010) | Q | 242, 243 |
| | 3.9 | | Raventos-Duran et al. (2010) | Q | 244 |
| | $1.6\times10^1$ | | Raventos-Duran et al. (2010) | Q | 245 |
| | 9.2 | | Hilal et al. (2008) | Q | |
| | 1.8 | | Modarresi et al. (2007) | Q | 67 |
| | 3.1 | | Yaffe et al. (2003) | Q | 248, 249 |
| | $1.2\times10^{-2}$ | | Katritzky et al. (1998) | Q | |
| | $4.6\times10^1$ | | Nirmalakhandan et al. (1997) | Q | |
| | 1.5 | | Duchowicz et al. (2020) | ? | 185, 21 |
| | 1.4 | | Yaws (1999) | ? | 21 |
| 1-hydroxy-3,4-dimethylbenzene | $2.6\times10^1$ | 7200 | Brockbank (2013) | L | 1 |
| $C_8H_{10}O$ | $2.4\times10^1$ | 7100 | Dohnal and Fenclová (1995) | M | |
| (3,4-xylenol; 3,4-dimethylphenol) | 8.2 | | HSDB (2015) | V | |
| [95-65-8] | $4.6\times10^1$ | | Mackay et al. (2006c) | V | |
| YCOXTKKNXUZSKD-UHFFFAOYSA-N | $4.7\times10^1$ | | Mackay et al. (1995) | V | |
| | $4.7\times10^1$ | | Shiu et al. (1994) | V | |
| | $1.1\times10^2$ | | Leuenberger et al. (1985) | V | 416 |
| | $2.4\times10^1$ | | Abraham et al. (1994a) | R | |
| | $1.1\times10^1$ | | Yaws (2003) | X | 258 |
| | $1.8\times10^1$ | | Dupeux et al. (2022) | Q | 259 |
| | 7.9 | | Keshavarz et al. (2022) | Q | |
| | $1.2\times10^1$ | | Duchowicz et al. (2020) | Q | 299 |
| | $1.1\times10^1$ | | Gharagheizi et al. (2012) | Q | |
| | 6.2 | | Raventos-Duran et al. (2010) | Q | 242, 243 |
| | 7.8 | | Raventos-Duran et al. (2010) | Q | 244 |
| | $1.6\times10^1$ | | Raventos-Duran et al. (2010) | Q | 245 |
| | 4.4 | | Hilal et al. (2008) | Q | |
| | $1.2\times10^1$ | | Modarresi et al. (2007) | Q | 67 |
| | $2.4\times10^1$ | | Yaffe et al. (2003) | Q | 248, 249 |
| | $2.1\times10^1$ | | English and Carroll (2001) | Q | 230, 260 |
| | $4.6\times10^1$ | | Nirmalakhandan et al. (1997) | Q | |
| | $2.4\times10^1$ | | Duchowicz et al. (2020) | ? | 185, 21 |





Table A3.2: Alcohols (ROH) (...continued)

| Substance / Formula / (Trivial Name) / [CAS Registry Number] / InChIKey | $H_s^{cp}$ (at $T^\ominus$) $\left[\dfrac{\text{mol}}{\text{m}^3\,\text{Pa}}\right]$ | $\dfrac{\mathrm{d}\ln H_s^{cp}}{\mathrm{d}(1/T)}$ [K] | Reference | Type | Note |
|---|---|---|---|---|---|
| 1-hydroxy-3,5-dimethylbenzene | $1.5\times10^1$ | 7600 | Brockbank (2013) | L | 1 |
| $C_8H_{10}O$ | $1.6\times10^1$ | 6900 | Dohnal and Fenclová (1995) | M | |
| (3,5-xylenol; 3,5-dimethylphenol) | 7.6 | | HSDB (2015) | V | |
| [108-68-9] | $2.8\times10^1$ | | Mackay et al. (2006c) | V | |
| TUAMRELNJMMDMT-UHFFFAOYSA-N | $3.1\times10^1$ | | Mackay et al. (1995) | V | |
| | $2.5\times10^1$ | | Shiu et al. (1994) | V | |
| | $6.2\times10^1$ | | Leuenberger et al. (1985) | V | 416 |
| | $1.6\times10^1$ | | Abraham et al. (1994a) | R | |
| | 7.3 | | Yaws (2003) | X | 258 |
| | $1.2\times10^1$ | | Dupeux et al. (2022) | Q | 259 |
| | 7.9 | | Keshavarz et al. (2022) | Q | |
| | $1.2\times10^1$ | | Duchowicz et al. (2020) | Q | 184 |
| | 6.2 | | Raventos-Duran et al. (2010) | Q | 242, 243 |
| | 4.9 | | Raventos-Duran et al. (2010) | Q | 244 |
| | $1.6\times10^1$ | | Raventos-Duran et al. (2010) | Q | 245 |
| | 3.2 | | Hilal et al. (2008) | Q | |
| | $1.1\times10^1$ | | Modarresi et al. (2007) | Q | 67 |
| | $1.6\times10^1$ | | Yaffe et al. (2003) | Q | 248, 249 |
| | $1.7\times10^1$ | | English and Carroll (2001) | Q | 230, 231 |
| | $4.6\times10^1$ | | Nirmalakhandan et al. (1997) | Q | |
| | $1.6\times10^1$ | | Duchowicz et al. (2020) | ? | 185, 21 |
| 2-methylbenzenemethanol | $2.2\times10^1$ | | Wang et al. (2017) | Q | 80, 238 |
| $C_8H_{10}O$ | $5.6\times10^1$ | | Wang et al. (2017) | Q | 80, 239 |
| [89-95-2] | $1.7\times10^1$ | | Wang et al. (2017) | Q | 80, 240 |
| XPNGNIFUDRPBFJ-UHFFFAOYSA-N | | | | | |
| 3-methylbenzenemethanol | $2.2\times10^1$ | | Wang et al. (2017) | Q | 80, 238 |
| $C_8H_{10}O$ | $4.5\times10^1$ | | Wang et al. (2017) | Q | 80, 239 |
| [587-03-1] | $3.4\times10^1$ | | Wang et al. (2017) | Q | 80, 240 |
| JJCKHVUTVOPLBV-UHFFFAOYSA-N | | | | | |
| 4-methylbenzenemethanol | 9.0 | | HSDB (2015) | V | |
| $C_8H_{10}O$ | $2.2\times10^1$ | | Wang et al. (2017) | Q | 80, 238 |
| [589-18-4] | $5.3\times10^1$ | | Wang et al. (2017) | Q | 80, 239 |
| KMTDMTZBNYGUNX-UHFFFAOYSA-N | $4.6\times10^1$ | | Wang et al. (2017) | Q | 80, 240 |
| $\alpha$-methylbenzyl alcohol | $3.4\times10^1$ | | HSDB (2015) | Q | 99 |
| $C_8H_{10}O$ | | | | | |
| [98-85-1] | | | | | |
| WAPNOHKVXSQRPX-UHFFFAOYSA-N | | | | | |
| 1-hydroxy-2-ethylbenzene | 2.1 | | Duchowicz et al. (2020) | V | 186 |
| $C_8H_{10}O$ | 2.1 | | HSDB (2015) | V | |
| (2-ethylphenol) | 5.6 | | Mackay et al. (2006c) | V | |
| [90-00-6] | $2.4\times10^1$ | | Duchowicz et al. (2020) | Q | |
| IXQGCWUGDFDQMF-UHFFFAOYSA-N | $3.0\times10^1$ | | Wang et al. (2017) | Q | 80, 238 |
| | 8.1 | | Wang et al. (2017) | Q | 80, 239 |
| | 7.3 | | Wang et al. (2017) | Q | 80, 240 |
| | 6.2 | | Raventos-Duran et al. (2010) | Q | 242, 243 |





Table A3.2: Alcohols (ROH) (...continued)

| Substance / Formula / (Trivial Name) / [CAS Registry Number] / InChIKey | $H_s^{cp}$ (at $T^{\ominus}$) $\left[\dfrac{\text{mol}}{\text{m}^3\,\text{Pa}}\right]$ | $\dfrac{\text{d}\ln H_s^{cp}}{\text{d}(1/T)}$ [K] | Reference | Type | Note |
|---|---|---|---|---|---|
| | 4.9 | | Raventos-Duran et al. (2010) | Q | 244 |
| | $1.2\times10^1$ | | Raventos-Duran et al. (2010) | Q | 245 |
| | 1.6 | | Modarresi et al. (2007) | Q | 67 |
| | 2.2 | | Yaffe et al. (2003) | Q | 248, 249 |
| | 3.1 | | Katritzky et al. (1998) | Q | |
| 1-hydroxy-3-ethylbenzene | 4.9 | | Karl et al. (2003) | M | |
| $C_8H_{10}O$ | $1.6\times10^1$ | | Abraham et al. (1994a) | R | |
| (3-ethylphenol) | 7.9 | | Keshavarz et al. (2022) | Q | |
| [620-17-7] | $2.4\times10^1$ | | Duchowicz et al. (2020) | Q | 184 |
| HMNKTRSOROOSPP-UHFFFAOYSA-N | 9.0 | | HSDB (2015) | Q | 99 |
| | 6.2 | | Raventos-Duran et al. (2010) | Q | 242, 243 |
| | 4.9 | | Raventos-Duran et al. (2010) | Q | 244 |
| | $1.2\times10^1$ | | Raventos-Duran et al. (2010) | Q | 245 |
| | 3.4 | | Hilal et al. (2008) | Q | |
| | $2.1\times10^1$ | | Modarresi et al. (2007) | Q | 67 |
| | $1.5\times10^1$ | | Yaffe et al. (2003) | Q | 248, 249 |
| | $1.8\times10^1$ | | English and Carroll (2001) | Q | 230, 231 |
| | $5.4\times10^1$ | | Nirmalakhandan et al. (1997) | Q | |
| | $1.6\times10^1$ | | Duchowicz et al. (2020) | ? | 185, 21 |
| 1-hydroxy-4-ethylbenzene | 8.2 | | HSDB (2015) | V | |
| $C_8H_{10}O$ | $2.1\times10^1$ | | Mackay et al. (2006c) | V | |
| (4-ethylphenol) | $1.3\times10^1$ | | Abraham et al. (1994a) | R | |
| [123-07-9] | 7.9 | | Keshavarz et al. (2022) | Q | |
| HXDOZKJGKXYMEW-UHFFFAOYSA-N | $2.4\times10^1$ | | Duchowicz et al. (2020) | Q | 299 |
| | 6.2 | | Raventos-Duran et al. (2010) | Q | 271, 243 |
| | 6.2 | | Raventos-Duran et al. (2010) | Q | 244 |
| | $1.2\times10^1$ | | Raventos-Duran et al. (2010) | Q | 245 |
| | 3.8 | | Hilal et al. (2008) | Q | |
| | 9.4 | | Modarresi et al. (2007) | Q | 67 |
| | $1.6\times10^1$ | | English and Carroll (2001) | Q | 230, 231 |
| | $5.4\times10^1$ | | Nirmalakhandan et al. (1997) | Q | |
| | $1.3\times10^1$ | | Duchowicz et al. (2020) | ? | 185, 21 |
| 2,3,4-trimethylphenol | $1.3\times10^1$ | | Wang et al. (2017) | Q | 80, 238 |
| $C_9H_{12}O$ | $1.8\times10^1$ | | Wang et al. (2017) | Q | 80, 239 |
| [526-85-2] | 8.9 | | Wang et al. (2017) | Q | 80, 240 |
| XRUGBBIQLIVCSI-UHFFFAOYSA-N | | | | | |
| 2,3,5-trimethylphenol | $1.2\times10^1$ | | Mackay et al. (2006c) | V | |
| $C_9H_{12}O$ | $1.2\times10^1$ | | Mackay et al. (1995) | V | |
| [697-82-5] | | | | | |
| OGRAOKJKVGDSFR-UHFFFAOYSA-N | | | | | |



Table A3.2: Alcohols (ROH) (...continued)

| Substance<br>Formula<br>(Trivial Name)<br>[CAS Registry Number]<br>InChIKey | $H_s^{cp}$<br>(at $T^\ominus$)<br>$\left[\dfrac{\text{mol}}{\text{m}^3\,\text{Pa}}\right]$ | $\dfrac{\text{d}\ln H_s^{cp}}{\text{d}(1/T)}$<br><br>[K] | Reference | Type | Note |
|---|---|---|---|---|---|
| 2,3,6-trimethylphenol<br>$C_9H_{12}O$<br>[2416-94-6]<br>QQOMQLYQAXGHSU-UHFFFAOYSA-N | 2.5<br>2.5<br>1.6<br>2.5<br>$1.1\times10^1$<br>1.3<br>$1.1\times10^1$<br>$4.1\times10^{-1}$ | | Duchowicz et al. (2020)<br>HSDB (2015)<br>Duchowicz et al. (2020)<br>Wang et al. (2017)<br>Wang et al. (2017)<br>Wang et al. (2017)<br>Hilal et al. (2008)<br>Modarresi et al. (2007) | V<br>V<br>Q<br>Q<br>Q<br>Q<br>Q<br>Q | 186<br><br><br>80, 238<br>80, 239<br>80, 240<br><br>67 |
| 2,4,6-trimethylphenol<br>$C_9H_{12}O$<br>[527-60-6]<br>BPRYUXCVCCNUFE-UHFFFAOYSA-N | 3.8<br>3.2<br>1.3<br>1.4<br>1.6<br>2.5<br>8.9<br>1.2<br>4.9<br>3.1<br>$1.2\times10^1$<br>9.2<br>1.2 | | Duchowicz et al. (2020)<br>HSDB (2015)<br>Mackay et al. (2006c)<br>Mackay et al. (1995)<br>Duchowicz et al. (2020)<br>Wang et al. (2017)<br>Wang et al. (2017)<br>Wang et al. (2017)<br>Raventos-Duran et al. (2010)<br>Raventos-Duran et al. (2010)<br>Raventos-Duran et al. (2010)<br>Hilal et al. (2008)<br>Modarresi et al. (2007) | V<br>V<br>V<br>V<br>Q<br>Q<br>Q<br>Q<br>Q<br>Q<br>Q<br>Q<br>Q | 186<br><br><br><br><br>80, 238<br>80, 239<br>80, 240<br>242, 243<br>244<br>245<br><br>67 |
| 3,4,5-trimethylphenol<br>$C_9H_{12}O$<br>[527-54-8]<br>FDQQNNZKEJIHMS-UHFFFAOYSA-N | $3.4\times10^1$<br>$3.8\times10^1$ | | Mackay et al. (2006c)<br>Mackay et al. (1995) | V<br>V | |
| 1-hydroxy-4-propylbenzene<br>$C_9H_{12}O$<br>(4-propylphenol)<br>[645-56-7]<br>KLSLBUSXWBJMEC-UHFFFAOYSA-N | 1.7<br>8.6<br>$1.1\times10^1$<br>$2.5\times10^1$<br>4.9<br>3.9<br>9.9<br>3.1<br>$1.0\times10^1$<br>8.8<br>$1.2\times10^1$<br>$4.3\times10^1$<br>8.7 | | Mackay et al. (2006c)<br>Abraham et al. (1994a)<br>Keshavarz et al. (2022)<br>Duchowicz et al. (2020)<br>Raventos-Duran et al. (2010)<br>Raventos-Duran et al. (2010)<br>Raventos-Duran et al. (2010)<br>Hilal et al. (2008)<br>Modarresi et al. (2007)<br>Yaffe et al. (2003)<br>English and Carroll (2001)<br>Nirmalakhandan et al. (1997)<br>Duchowicz et al. (2020) | V<br>R<br>Q<br>Q<br>Q<br>Q<br>Q<br>Q<br>Q<br>Q<br>Q<br>Q<br>? | <br><br><br><br>271, 243<br>244<br>245<br><br>67<br>248, 249<br>230, 260<br><br>185, 21 |
| 2-(1-methylethyl)-phenol<br>$C_9H_{12}O$<br>[88-69-7]<br>CRBJBYGJVIBWIY-UHFFFAOYSA-N | 2.6<br>$2.8\times10^1$<br>3.9<br>4.5<br>2.8 | | Mackay et al. (2006c)<br>Wang et al. (2017)<br>Wang et al. (2017)<br>Wang et al. (2017)<br>Hilal et al. (2008) | V<br>Q<br>Q<br>Q<br>Q | <br>80, 238<br>80, 239<br>80, 240<br> |



Table A3.2: Alcohols (ROH) (. . . continued)

| Substance<br>Formula<br>(Trivial Name)<br>[CAS Registry Number]<br>InChIKey | $H_s^{cp}$<br>(at $T^\ominus$)<br>$\left[\dfrac{\mathrm{mol}}{\mathrm{m^3\,Pa}}\right]$ | $\dfrac{\mathrm{d}\ln H_s^{cp}}{\mathrm{d}(1/T)}$<br><br>[K] | Reference | Type | Note |
|---|---|---|---|---|---|
| 2-phenylisopropanol<br>$C_9H_{12}O$<br>[617-94-7]<br>BDCFWIDZNLCTMF-UHFFFAOYSA-N | $2.0\times10^1$<br>$1.4\times10^1$<br>$1.2\times10^1$<br>$2.6\times10^1$ | | Wang et al. (2017)<br>Wang et al. (2017)<br>Wang et al. (2017)<br>HSDB (2015) | Q<br>Q<br>Q<br>Q | 80, 238<br>80, 239<br>80, 240<br>99 |
| 2-ethyl-3-methylphenol<br>$C_9H_{12}O$<br>[6161-62-2]<br>OCKYMBMCPOAFLL-UHFFFAOYSA-N | $1.8\times10^1$<br>8.9<br>5.8 | | Wang et al. (2017)<br>Wang et al. (2017)<br>Wang et al. (2017) | Q<br>Q<br>Q | 80, 238<br>80, 239<br>80, 240 |
| 2-ethyl-5-methylphenol<br>$C_9H_{12}O$<br>[1687-61-2]<br>LTRVUFFOMIUCPJ-UHFFFAOYSA-N | $1.8\times10^1$<br>8.0<br>4.9 | | Wang et al. (2017)<br>Wang et al. (2017)<br>Wang et al. (2017) | Q<br>Q<br>Q | 80, 238<br>80, 239<br>80, 240 |
| 2-ethyl-6-methylphenol<br>$C_9H_{12}O$<br>[1687-64-5]<br>CIRRFAQIWQFQSS-UHFFFAOYSA-N | 3.3<br>6.2<br>$6.9\times10^{-1}$ | | Wang et al. (2017)<br>Wang et al. (2017)<br>Wang et al. (2017) | Q<br>Q<br>Q | 80, 238<br>80, 239<br>80, 240 |
| 5-ethyl-3-methylphenol<br>$C_9H_{12}O$<br>[698-71-5]<br>XTCHLXABLZQNNN-UHFFFAOYSA-N | $1.5\times10^1$<br>$1.3\times10^1$<br>4.9<br>3.9<br>9.9<br>2.9<br>$1.8\times10^1$ | | Duchowicz et al. (2020)<br>Duchowicz et al. (2020)<br>Raventos-Duran et al. (2010)<br>Raventos-Duran et al. (2010)<br>Raventos-Duran et al. (2010)<br>Hilal et al. (2008)<br>Modarresi et al. (2007) | V<br>Q<br>Q<br>Q<br>Q<br>Q<br>Q | 186<br><br>242, 243<br>244<br>245<br><br>67 |
| 1-(2-methylphenyl)ethanol<br>$C_9H_{12}O$<br>[7287-82-3]<br>SDCBYRLJYGORNK-UHFFFAOYSA-N | $2.1\times10^1$<br>$2.9\times10^1$<br>$1.6\times10^1$ | | Wang et al. (2017)<br>Wang et al. (2017)<br>Wang et al. (2017) | Q<br>Q<br>Q | 80, 238<br>80, 239<br>80, 240 |
| 2-propylphenol<br>$C_9H_{12}O$<br>[644-35-9]<br>LCHYEKKJCUJAKN-UHFFFAOYSA-N | $2.7\times10^1$<br>5.6<br>5.0 | | Wang et al. (2017)<br>Wang et al. (2017)<br>Wang et al. (2017) | Q<br>Q<br>Q | 80, 238<br>80, 239<br>80, 240 |
| 1-phenyl-1-propanol<br>$C_9H_{12}O$<br>[93-54-9]<br>DYUQAZSOFZSPHD-UHFFFAOYSA-N | $2.8\times10^1$<br>$2.3\times10^1$<br>9.8 | | Wang et al. (2017)<br>Wang et al. (2017)<br>Wang et al. (2017) | Q<br>Q<br>Q | 80, 238<br>80, 239<br>80, 240 |
| 2,3-dimethylbenzyl alcohol<br>$C_9H_{12}O$<br>[13651-14-4]<br>ZQQIVMXQYUZKIQ-UHFFFAOYSA-N | $1.3\times10^1$<br>$7.4\times10^1$<br>$2.3\times10^1$ | | Wang et al. (2017)<br>Wang et al. (2017)<br>Wang et al. (2017) | Q<br>Q<br>Q | 80, 238<br>80, 239<br>80, 240 |





Table A3.2: Alcohols (ROH) (...continued)

| Substance Formula (Trivial Name) [CAS Registry Number] InChIKey | $H_s^{cp}$ (at $T^{\ominus}$) $\left[\dfrac{\mathrm{mol}}{\mathrm{m^3\,Pa}}\right]$ | $\dfrac{\mathrm{d}\ln H_s^{cp}}{\mathrm{d}(1/T)}$ [K] | Reference | Type | Note |
|---|---|---|---|---|---|
| 3,4-dimethylbenzyl alcohol $C_9H_{12}O$ [6966-10-5] OKGZCXPDJKKZAP-UHFFFAOYSA-N | $1.3\times10^1$ $6.2\times10^1$ $5.5\times10^1$ | | Wang et al. (2017) Wang et al. (2017) Wang et al. (2017) | Q Q Q | 80, 238 80, 239 80, 240 |
| 3,5-dimethylbenzyl alcohol $C_9H_{12}O$ [27129-87-9] IQWWTJDRVBWBEL-UHFFFAOYSA-N | $1.3\times10^1$ $3.7\times10^1$ $4.0\times10^1$ | | Wang et al. (2017) Wang et al. (2017) Wang et al. (2017) | Q Q Q | 80, 238 80, 239 80, 240 |
| 2-(1,1-dimethylethyl)-phenol $C_{10}H_{14}O$ [88-18-6] WJQOZHYUIDYNHM-UHFFFAOYSA-N | $3.9\times10^{-1}$ 4.4 7.0 | | Duchowicz et al. (2020) Duchowicz et al. (2020) HSDB (2015) | V Q Q | 186  99 |
| 3-(1,1-dimethylethyl)-phenol $C_{10}H_{14}O$ [585-34-2] CYEKUDPFXBLGHH-UHFFFAOYSA-N | 5.1 4.4 | | Duchowicz et al. (2020) Duchowicz et al. (2020) | V Q | 186 |
| 2-(1-methylpropyl)phenol $C_{10}H_{14}O$ [89-72-5] NGFPWHGISWUQOI-UHFFFAOYSA-N | 4.7 | | HSDB (2015) | Q | 99 |
| 4-(1-methylpropyl)-phenol $C_{10}H_{14}O$ (4-*sec*-butylphenol) [99-71-8] ZUTYZAFDFLLILI-UHFFFAOYSA-N | 3.6 4.3 | | Mackay et al. (2006c) Mackay et al. (1995) | V V | |
| 4-*tert*-butylphenol $C_{10}H_{14}O$ [98-54-4] QHPQWRBYOIRBIT-UHFFFAOYSA-N | 8.9 $1.6\times10^1$ $2.1\times10^1$ $1.4\times10^1$ 4.4 2.1 4.8 8.8 6.7 $2.5\times10^1$ $2.4\times10^1$ 5.2 2.7 8.3 $1.5\times10^{-1}$ 8.8 | 7700 | Parsons et al. (1972) Mackay et al. (2006c) Mackay et al. (1995) Keshavarz et al. (2022) Duchowicz et al. (2020) Hilal et al. (2008) Modarresi et al. (2007) Yaffe et al. (2003) English and Carroll (2001) Katritzky et al. (1998) Nirmalakhandan et al. (1997) Suzuki et al. (1992) Nirmalakhandan and Speece (1988) Duchowicz et al. (2020) Betterton (1992) Abraham et al. (1990) | M V V Q Q Q Q Q Q Q Q Q Q ? ? ? | 417    67 248, 249 230, 231   232  185, 21 421 |



Table A3.2: Alcohols (ROH) (...continued)

| Substance Formula (Trivial Name) [CAS Registry Number] InChIKey | $H_s^{cp}$ (at $T^\ominus$) $\left[\dfrac{\text{mol}}{\text{m}^3\,\text{Pa}}\right]$ | $\dfrac{\text{d}\ln H_s^{cp}}{\text{d}(1/T)}$ [K] | Reference | Type | Note |
|---|---|---|---|---|---|
| 2-methyl-5-(1-methylethyl)-phenol $C_{10}H_{14}O$ (carvacrol) [499-75-2] RECUKUPTGUEGMW-UHFFFAOYSA-N | 1.5 2.4 | 9300 | Martins et al. (2017) van Roon et al. (2005) | V V | 315 |
| 5-methyl-2-(1-methylethyl)-phenol $C_{10}H_{14}O$ (thymol) [89-83-8] MGSRCZKZVOBKFT-UHFFFAOYSA-N | $2.0\times10^1$ 1.5 3.0 5.1 3.1 2.0 7.8 2.8 $7.1\times10^{-1}$ | 9300 | Duchowicz et al. (2020) Martins et al. (2017) van Roon et al. (2005) Duchowicz et al. (2020) Raventos-Duran et al. (2010) Raventos-Duran et al. (2010) Raventos-Duran et al. (2010) Hilal et al. (2008) Modarresi et al. (2007) | V V V Q Q Q Q Q Q | 186 315 242, 243 244 245 67 |
| 1-(3,5-dimethylphenyl)ethanol $C_{10}H_{14}O$ [5379-18-0] RHBAJFPGUNNLFA-UHFFFAOYSA-N | $1.2\times10^1$ $1.8\times10^1$ $1.4\times10^1$ | | Wang et al. (2017) Wang et al. (2017) Wang et al. (2017) | Q Q Q | 80, 238 80, 239 80, 240 |
| 6-ethyl-2,4-xylenol $C_{10}H_{14}O$ [2219-79-6] MXHAHSBTOVFDBK-UHFFFAOYSA-N | 2.0 5.8 $6.0\times10^{-1}$ | | Wang et al. (2017) Wang et al. (2017) Wang et al. (2017) | Q Q Q | 80, 238 80, 239 80, 240 |
| 2-(1,1-dimethylethyl)-4-methylphenol $C_{11}H_{16}O$ [2409-55-4] IKEHOXWJQXIQAG-UHFFFAOYSA-N | 6.6 | | HSDB (2015) | Q | 99 |
| 4-(1,1-dimethylpropyl)phenol $C_{11}H_{16}O$ [80-46-6] NRZWYNLTFLDQQX-UHFFFAOYSA-N | 4.9 | | HSDB (2015) | V | |
| MCM:DE35TOH $C_{11}H_{16}O$ OZZOZHVUMWFILP-UHFFFAOYSA-N | $1.0\times10^1$ $1.6\times10^1$ $1.4\times10^1$ | | Wang et al. (2017) Wang et al. (2017) Wang et al. (2017) | Q Q Q | 80, 238 80, 239 80, 240 |
| MCM:DEMPHOH $C_{11}H_{16}O$ VUDWBUZLGAXSDX-UHFFFAOYSA-N | 1.8 3.8 $6.3\times10^{-1}$ | | Wang et al. (2017) Wang et al. (2017) Wang et al. (2017) | Q Q Q | 80, 238 80, 239 80, 240 |
| josenol $C_{11}H_{14}O$ (2-methyl-3-(4-methylphenyl)-2-propen-1-ol) [56138-10-4] WWHATUWNHYNMNW-UHFFFAOYSA-N | $4.5\times10^1$ | | Dupeux et al. (2022) | Q | 259 |



Table A3.2: Alcohols (ROH) (. . . continued)

| Substance<br>Formula<br>(Trivial Name)<br>[CAS Registry Number]<br>InChIKey | $H_s^{cp}$<br>(at $T^\ominus$)<br>$\left[\dfrac{\text{mol}}{\text{m}^3\,\text{Pa}}\right]$ | $\dfrac{\text{d}\ln H_s^{cp}}{\text{d}(1/T)}$<br><br>[K] | Reference | Type | Note |
|---|---|---|---|---|---|
| lilyflore<br>$C_{12}H_{16}O$<br>[285977-85-7]<br>UWSPWQQZFOSTHS-UHFFFAOYSA-N | $1.3\times10^1$ | | Dupeux et al. (2022) | Q | 259 |
| benzhydrol<br>$C_{13}H_{12}O$<br>[91-01-0]<br>QILSFLSDHQAZET-UHFFFAOYSA-N | $3.8\times10^2$ | | Ebert et al. (2023) | ? | 318 |
| 1-hydroxy-4-octylbenzene<br>$C_{14}H_{22}O$<br>(4-octylphenol)<br>[1806-26-4]<br>NTDQQZYCCIDJRK-UHFFFAOYSA-N | 1.3<br>2.0 | | Mackay et al. (2006c)<br>Mackay et al. (1995) | V<br>V | |
| 4-(1,1,3,3-tetramethylbutyl)-phenol<br>$C_{14}H_{22}O$<br>(*p-tert*-octylphenol)<br>[140-66-9]<br>ISAVYTVYFVQUDY-UHFFFAOYSA-N | 2.3<br>1.4<br>2.2<br>2.3<br>$1.0\times10^1$<br>1.8 | 9000 | Xie et al. (2004)<br>HSDB (2015)<br>Zhang et al. (2010)<br>Zhang et al. (2010)<br>Zhang et al. (2010)<br>Zhang et al. (2010) | M<br>Q<br>Q<br>Q<br>Q<br>Q | <br>99<br>287, 288<br>287, 289<br>287, 290<br>287, 291 |
| 1-hydroxy-4-nonylbenzene<br>$C_{15}H_{24}O$<br>(4-nonylphenol)<br>[104-40-5]<br>IGFHQQFPSIBGKE-UHFFFAOYSA-N | $2.9\times10^{-1}$<br>$2.9\times10^{-1}$<br>$3.6\times10^{-1}$<br>$6.4\times10^{-1}$<br>$2.8\times10^1$ | | Duchowicz et al. (2020)<br>HSDB (2015)<br>Mackay et al. (2006c)<br>Mackay et al. (1995)<br>Duchowicz et al. (2020) | V<br>V<br>V<br>V<br>Q | 186 |
| 2,6-bis(1,1-dimethylethyl)-4-methylphenol<br>$C_{15}H_{24}O$<br>(butylated hydroxytoluene; BHT)<br>[128-37-0]<br>NLZUEZXRPGMBCV-UHFFFAOYSA-N | $2.9\times10^{-3}$ | | Yoshida et al. (1983) | V | |
| 4-(3',5'-dimethyl-3'-heptyl)-phenol(+)<br>$C_{15}H_{24}O$<br>RYIHVIPUIXFRNI-IUODEOHRSA-N | 2.9 | 8700 | Xie et al. (2004) | M | |
| 4-(3',5'-dimethyl-3'-heptyl)-phenol(-)<br>$C_{15}H_{24}O$<br>RYIHVIPUIXFRNI-SWLSCSKDSA-N | 3.3 | 8600 | Xie et al. (2004) | M | |





Table A3.2: Alcohols (ROH) (...continued)

| Substance Formula (Trivial Name) [CAS Registry Number] InChIKey | $H_s^{cp}$ (at $T^\ominus$) $\left[\dfrac{\mathrm{mol}}{\mathrm{m^3\,Pa}}\right]$ | $\dfrac{\mathrm{d}\ln H_s^{cp}}{\mathrm{d}(1/T)}$ [K] | Reference | Type | Note |
|---|---|---|---|---|---|
| 2-phenylethanol | $1.7\times10^1$ | 7200 | Brockbank (2013) | L | 1 |
| $C_8H_{10}O$ | $>3.7\times10^1$ | | Altschuh et al. (1999) | M | |
| [60-12-8] | $6.6\times10^1$ | | HSDB (2015) | V | |
| WRMNZCZEMHIOCP-UHFFFAOYSA-N | $3.9\times10^1$ | | Abraham et al. (1994a) | R | |
| | 3.5 | | Yaws (2003) | X | 258 |
| | $1.6\times10^1$ | | Dupeux et al. (2022) | Q | 259 |
| | $3.0\times10^1$ | | Keshavarz et al. (2022) | Q | |
| | $2.0\times10^{-3}$ | | Abney (2021) | Q | 399 |
| | 8.5 | | Duchowicz et al. (2020) | Q | 184 |
| | 8.5 | | Wang et al. (2017) | Q | 80, 238 |
| | $3.9\times10^1$ | | Wang et al. (2017) | Q | 80, 239 |
| | $3.9\times10^1$ | | Wang et al. (2017) | Q | 80, 240 |
| | $7.8\times10^1$ | | Raventos-Duran et al. (2010) | Q | 242, 243 |
| | $2.5\times10^1$ | | Raventos-Duran et al. (2010) | Q | 244 |
| | $3.1\times10^1$ | | Raventos-Duran et al. (2010) | Q | 245 |
| | $1.9\times10^1$ | | Hilal et al. (2008) | Q | |
| | $3.2\times10^1$ | | Modarresi et al. (2007) | Q | 67 |
| | $2.4\times10^{-1}$ | | Emel'yanenko et al. (2007) | Q | 415 |
| | $2.4\times10^{-1}$ | | Hertel and Sommer (2005) | Q | 415 |
| | $3.9\times10^1$ | | Yaffe et al. (2003) | Q | 248, 249 |
| | $1.0\times10^1$ | | Yao et al. (2002) | Q | 229 |
| | $2.8\times10^1$ | | English and Carroll (2001) | Q | 230, 274 |
| | $5.4\times10^1$ | | Katritzky et al. (1998) | Q | |
| | $5.3\times10^1$ | | Nirmalakhandan et al. (1997) | Q | |
| | $3.9\times10^1$ | | Duchowicz et al. (2020) | ? | 185, 21 |
| | 3.5 | | Yaws (1999) | ? | 21 |
| 3-phenyl-1-propanol | $>1.8\times10^2$ | | Altschuh et al. (1999) | M | |
| $C_9H_{12}O$ | $4.8\times10^1$ | | Abraham et al. (1994a) | R | |
| [122-97-4] | $4.1\times10^1$ | | Keshavarz et al. (2022) | Q | |
| VAJVDSVGBWFCLW-UHFFFAOYSA-N | 8.8 | | Duchowicz et al. (2020) | Q | 299 |
| | $1.4\times10^1$ | | Hilal et al. (2008) | Q | |
| | $2.0\times10^1$ | | Modarresi et al. (2007) | Q | 67 |
| | $3.4\times10^1$ | | English and Carroll (2001) | Q | 230, 231 |
| | $4.2\times10^1$ | | Nirmalakhandan et al. (1997) | Q | |
| | $4.9\times10^1$ | | Duchowicz et al. (2020) | ? | 185, 21 |
| 4-phenyl-1-butanol | $>6.7$ | | Altschuh et al. (1999) | M | |
| $C_{10}H_{14}O$ | $1.2\times10^1$ | | Hilal et al. (2008) | Q | |
| [3360-41-6] | | | | | |
| LDZLXQFDGRCELX-UHFFFAOYSA-N | | | | | |
| 1-naphthalenol | $1.6\times10^2$ | | HSDB (2015) | V | |
| $C_{10}H_8O$ | $2.9\times10^1$ | | Mackay et al. (2006c) | V | |
| (1-naphthol) | $1.7\times10^2$ | | Abraham et al. (1994a) | R | |
| [90-15-3] | $2.1\times10^2$ | | Keshavarz et al. (2022) | Q | |
| KJCVRFUGPWSIIH-UHFFFAOYSA-N | $1.7\times10^2$ | | Duchowicz et al. (2020) | Q | |
| | $6.9\times10^1$ | | Hilal et al. (2008) | Q | |
| | $8.4\times10^1$ | | Modarresi et al. (2007) | Q | 67 |





Table A3.2: Alcohols (ROH) (...continued)

| Substance Formula (Trivial Name) [CAS Registry Number] InChIKey | $H_s^{cp}$ (at $T^\ominus$) $\left[\dfrac{\mathrm{mol}}{\mathrm{m^3\,Pa}}\right]$ | $\dfrac{\mathrm{d}\ln H_s^{cp}}{\mathrm{d}(1/T)}$ [K] | Reference | Type | Note |
|---|---|---|---|---|---|
| | $4.5\times10^2$ | | English and Carroll (2001) | Q | 230, 231 |
| | $1.5\times10^3$ | | Nirmalakhandan et al. (1997) | Q | |
| | $1.6\times10^2$ | | Duchowicz et al. (2020) | ? | 185, 21 |
| 2-naphthalenol | $1.1\times10^2$ | | Mackay et al. (2006c) | V | |
| $C_{10}H_8O$ | $3.6\times10^2$ | | Abraham et al. (1994a) | R | |
| (2-naphthol) | $2.1\times10^2$ | | Keshavarz et al. (2022) | Q | |
| [135-19-3] | $1.7\times10^2$ | | Duchowicz et al. (2020) | Q | |
| JWAZRIHNYRIHIV-UHFFFAOYSA-N | $2.1\times10^2$ | | HSDB (2015) | Q | 99 |
| | $7.0\times10^1$ | | Hilal et al. (2008) | Q | |
| | $2.7\times10^2$ | | Modarresi et al. (2007) | Q | 67 |
| | | 7400 | Kühne et al. (2005) | Q | |
| | $5.8\times10^2$ | | English and Carroll (2001) | Q | 230, 231 |
| | $1.7\times10^3$ | | Nirmalakhandan et al. (1997) | Q | |
| | $3.6\times10^2$ | | Duchowicz et al. (2020) | ? | 185, 21 |
| | | 7200 | Kühne et al. (2005) | ? | |
| $o$-hydroxybiphenyl | 9.4 | | Duchowicz et al. (2020) | V | 186 |
| $C_{12}H_{10}O$ | 9.4 | | HSDB (2015) | V | |
| [90-43-7] | $2.9\times10^{-1}$ | | Mackay et al. (2006c) | V | |
| LLEMOWNGBBNAJR-UHFFFAOYSA-N | $2.1\times10^2$ | | Duchowicz et al. (2020) | Q | |
| | $3.1\times10^1$ | | Hilal et al. (2008) | Q | |
| | $1.1\times10^1$ | | Modarresi et al. (2007) | Q | 67 |
| $p$-hydroxybiphenyl | $1.6\times10^{-1}$ | | Mackay et al. (2006c) | V | |
| $C_{12}H_{10}O$ | $1.9\times10^2$ | | HSDB (2015) | Q | 99 |
| [92-69-3] | | | | | |
| YXVFYQXJAXKLAK-UHFFFAOYSA-N | | | | | |
| 2,4,6-tris(1,1-dimethylethyl)phenol | 1.0 | | Zhang et al. (2010) | Q | 287, 288 |
| $C_{18}H_{30}O$ | $5.6\times10^{-2}$ | | Zhang et al. (2010) | Q | 287, 289 |
| [732-26-3] | $3.3\times10^{-2}$ | | Zhang et al. (2010) | Q | 287, 290 |
| PFEFOYRSMXVNEL-UHFFFAOYSA-N | $5.3\times10^{-2}$ | | Zhang et al. (2010) | Q | 287, 291 |
| dehydroabietol | 8.4 | | Zhang et al. (2010) | Q | 287, 288 |
| $C_{20}H_{30}O$ | $1.8\times10^2$ | | Zhang et al. (2010) | Q | 287, 289 |
| [3772-55-2] | $2.4\times10^1$ | | Zhang et al. (2010) | Q | 287, 290 |
| WSKGRAGZAQRSED-IOJLRTSASA-N | $7.2\times10^{-1}$ | | Zhang et al. (2010) | Q | 287, 291 |
| 2,2'-methylenebis(6-(1,1-dimethylethyl)-4-methylphenol) $C_{23}H_{32}O_2$ [119-47-1] KGRVJHAUYBGFFP-UHFFFAOYSA-N | $1.2\times10^6$ | | HSDB (2015) | Q | 99 |
| 2,4-dinonylphenol | $1.5\times10^{-1}$ | | HSDB (2015) | Q | 99 |
| $C_{24}H_{42}O$ | $1.6\times10^{-1}$ | | Zhang et al. (2010) | Q | 287, 288 |
| [137-99-5] | $3.8\times10^{-1}$ | | Zhang et al. (2010) | Q | 287, 289 |
| FDAJTLLBHNHECW-UHFFFAOYSA-N | $7.0\times10^{-1}$ | | Zhang et al. (2010) | Q | 287, 290 |
| | $2.5\times10^{-1}$ | | Zhang et al. (2010) | Q | 287, 291 |



### A3.3 Polyols ($R(OH)_n$)

Table A3.3: Polyols ($R(OH)_n$)

| Substance Formula (Trivial Name) [CAS Registry Number] InChIKey | $H_s^{cp}$ (at $T^{\ominus}$) $\left[\dfrac{\mathrm{mol}}{\mathrm{m^3\,Pa}}\right]$ | $\dfrac{\mathrm{d}\ln H_s^{cp}}{\mathrm{d}(1/T)}$ [K] | Reference | Type | Note |
|---|---|---|---|---|---|
| methanediol | $9.9\times10^1$ | | Franco et al. (2021) | R | 422 |
| HOCH$_2$OH | $4.0\times10^3$ | | Mansfield (2020) | T | 423 |
| [463-57-0] | $7.1\times10^2$ | | Mansfield (2020) | T | 423 |
| CKFGINPQOCXMAZ-UHFFFAOYSA-N | | | | | |
| 1,1-ethanediol | $4.0\times10^1$ | | Mansfield (2020) | T | 423 |
| CH$_3$CH$_2$(OH)$_2$ | $1.4\times10^1$ | | Mansfield (2020) | T | 423 |
| [4433-56-1] | | | | | |
| AZHSSKPUVBVXLK-UHFFFAOYSA-N | | | | | |
| 1,2-ethanediol | $6.5\times10^3$ | | Burkholder et al. (2019) | L | |
| HO(CH$_2$)$_2$OH | $6.5\times10^3$ | | Burkholder et al. (2015) | L | |
| (ethylene glycol) | $4.0\times10^3$ | | Bone et al. (1983) | M | 12 |
| [107-21-1] | $1.6\times10^2$ | | Butler and Ramchandani (1935) | M | 424 |
| LYCAIKOWRPUZTN-UHFFFAOYSA-N | 4.7 | | HSDB (2015) | V | |
| | $6.5\times10^3$ | 8800 | Compernolle and Müller (2014b) | V | |
| | $5.0\times10^3$ | | Hwang et al. (1992) | V | |
| | $7.0\times10^3$ | | Yaws (2003) | X | 258 |
| | $7.0\times10^3$ | | Yaws (2003) | X | 237 |
| | $4.8\times10^3$ | | Dupeux et al. (2022) | Q | 259 |
| | $2.9\times10^2$ | | Keshavarz et al. (2022) | Q | |
| | $6.4\times10^2$ | | Duchowicz et al. (2020) | Q | 184 |
| | $3.3\times10^2$ | | Wang et al. (2017) | Q | 80, 238 |
| | $1.3\times10^3$ | | Wang et al. (2017) | Q | 80, 239 |
| | $6.3\times10^3$ | | Wang et al. (2017) | Q | 80, 240 |
| | $5.6\times10^2$ | | Olsen et al. (2016) | Q | 425 |
| | $4.0\times10^2$ | | Olsen et al. (2016) | Q | 426 |
| | $4.3\times10^2$ | | Olsen et al. (2016) | Q | 427 |
| | $1.3\times10^4$ | | Gharagheizi et al. (2012) | Q | |
| | $7.8\times10^3$ | | Raventos-Duran et al. (2010) | Q | 242, 243 |
| | $4.9\times10^2$ | | Raventos-Duran et al. (2010) | Q | 244 |
| | $7.8\times10^1$ | | Raventos-Duran et al. (2010) | Q | 245 |
| | $2.5\times10^3$ | | Gharagheizi et al. (2010) | Q | 246 |
| | $7.2\times10^2$ | | Hilal et al. (2008) | Q | |
| | $3.1\times10^3$ | | Modarresi et al. (2007) | Q | 67 |
| | $6.9\times10^3$ | | Yao et al. (2002) | Q | 229 |
| | $8.6\times10^2$ | | Katritzky et al. (1998) | Q | |
| | $1.6\times10^2$ | | Duchowicz et al. (2020) | ? | 185, 21 |
| | $5.2\times10^3$ | | Yaws (1999) | ? | 21 |





Table A3.3: Polyols ($R(OH)_n$) (... continued)

| Substance<br>Formula<br>(Trivial Name)<br>[CAS Registry Number]<br>InChIKey | $H_s^{cp}$<br>(at $T^\ominus$)<br>$\left[\dfrac{\mathrm{mol}}{\mathrm{m^3\,Pa}}\right]$ | $\dfrac{\mathrm{d\ln}H_s^{cp}}{\mathrm{d}(1/T)}$<br><br>[K] | Reference | Type | Note |
|---|---|---|---|---|---|
| 1,2-propanediol | $2.7\times10^3$ | | Burkholder et al. (2019) | L | |
| $C_3H_8O_2$ | $2.7\times10^3$ | | Burkholder et al. (2015) | L | |
| (propylene glycol) | $7.7\times10^2$ | | Duchowicz et al. (2020) | V | 186 |
| [57-55-6] | $7.6\times10^2$ | | HSDB (2015) | V | |
| DNIAPMSPPWPWPWGF-UHFFFAOYSA-N | $2.7\times10^3$ | 9500 | Compernolle and Müller (2014b) | V | |
| | $2.7\times10^3$ | | Yaws (2003) | X | 258 |
| | $2.7\times10^3$ | | Yaws (2003) | X | 237 |
| | $1.9\times10^3$ | | Dupeux et al. (2022) | Q | 259 |
| | $3.5\times10^2$ | | Duchowicz et al. (2020) | Q | |
| | $3.1\times10^2$ | | Wang et al. (2017) | Q | 80, 238 |
| | $1.0\times10^3$ | | Wang et al. (2017) | Q | 80, 239 |
| | $1.3\times10^3$ | | Wang et al. (2017) | Q | 80, 240 |
| | $3.9\times10^3$ | | Gharagheizi et al. (2012) | Q | |
| | $2.7\times10^3$ | | Gharagheizi et al. (2010) | Q | 246 |
| | | | Saxena and Hildemann (1996) | E | 401, 428 |
| 1,3-propanediol | $1.6\times10^4$ | | Burkholder et al. (2019) | L | |
| $C_3H_8O_2$ | $1.6\times10^4$ | | Burkholder et al. (2015) | L | |
| [504-63-2] | $9.1\times10^3$ | | Bone et al. (1983) | M | 12 |
| YPFDHNVEDLHUCE-UHFFFAOYSA-N | $1.6\times10^4$ | 9500 | Compernolle and Müller (2014b) | V | |
| | $7.4\times10^2$ | | Wang et al. (2017) | Q | 80, 238 |
| | $6.3\times10^3$ | | Wang et al. (2017) | Q | 80, 239 |
| | $4.8\times10^3$ | | Wang et al. (2017) | Q | 80, 240 |
| | $3.1\times10^3$ | | Raventos-Duran et al. (2010) | Q | 271, 243 |
| | $2.5\times10^3$ | | Raventos-Duran et al. (2010) | Q | 244 |
| | $6.2\times10^1$ | | Raventos-Duran et al. (2010) | Q | 245 |
| | $4.0\times10^2$ | | Hilal et al. (2008) | Q | |
| | $2.0\times10^3$ | | Modarresi et al. (2007) | Q | 67 |
| 1,2,3-propanetriol | $4.7\times10^6$ | | Burkholder et al. (2019) | L | |
| $C_3H_8O_3$ | $4.7\times10^6$ | | Burkholder et al. (2015) | L | |
| (glycerol) | $5.8\times10^2$ | | Butler and Ramchandani (1935) | M | 424 |
| [56-81-5] | $4.7\times10^6$ | 11000 | Compernolle and Müller (2014b) | V | |
| PEDCQBHIVMGVHV-UHFFFAOYSA-N | $5.0\times10^6$ | | Hwang et al. (1992) | V | |
| | $3.8\times10^2$ | | Keshavarz et al. (2022) | Q | |
| | $1.7\times10^5$ | | Duchowicz et al. (2020) | Q | |
| | | | Saxena and Hildemann (1996) | E | 401, 429 |
| | $5.7\times10^2$ | | Duchowicz et al. (2020) | ? | 185, 21 |
| 1,2-butanediol | $2.1\times10^3$ | | Burkholder et al. (2019) | L | 430 |
| $C_4H_{10}O_2$ | $2.1\times10^3$ | | Burkholder et al. (2015) | L | 431 |
| [584-03-2] | $>3.4\times10^2$ | | Altschuh et al. (1999) | M | |
| BMRWNKZVCUKKSR-UHFFFAOYSA-N | $1.7\times10^3$ | | Duchowicz et al. (2020) | V | 186 |
| | $2.1\times10^3$ | 9900 | Compernolle and Müller (2014b) | V | |
| | $4.4\times10^2$ | | Duchowicz et al. (2020) | Q | |
| | $2.9\times10^2$ | | Wang et al. (2017) | Q | 80, 238 |
| | $7.6\times10^2$ | | Wang et al. (2017) | Q | 80, 239 |
| | $4.8\times10^2$ | | Wang et al. (2017) | Q | 80, 240 |



Table A3.3: Polyols $(R(OH)_n)$ (... continued)

| Substance<br>Formula<br>(Trivial Name)<br>[CAS Registry Number]<br>InChIKey | $H_s^{cp}$<br>(at $T^{\ominus}$)<br>$\left[\dfrac{\mathrm{mol}}{\mathrm{m^3\,Pa}}\right]$ | $\dfrac{\mathrm{d}\ln H_s^{cp}}{\mathrm{d}(1/T)}$<br><br>[K] | Reference | Type | Note |
|---|---|---|---|---|---|
| 1,3-butanediol | $7.0\times10^3$ | | Burkholder et al. (2019) | L | |
| $C_4H_{10}O_2$ | $7.0\times10^3$ | | Burkholder et al. (2015) | L | |
| [107-88-0] | $4.1\times10^3$ | | Duchowicz et al. (2020) | V | 186 |
| PUPZLCDOIYMWBV-UHFFFAOYSA-N | $7.0\times10^3$ | 10000 | Compernolle and Müller (2014b) | V | |
| | $9.5\times10^2$ | | Duchowicz et al. (2020) | Q | |
| | $6.9\times10^2$ | | Wang et al. (2017) | Q | 80, 238 |
| | $4.0\times10^3$ | | Wang et al. (2017) | Q | 80, 239 |
| | $2.5\times10^3$ | | Wang et al. (2017) | Q | 80, 240 |
| | $4.9\times10^4$ | | Saxena and Hildemann (1996) | E | 401 |
| 1,4-butanediol | $3.5\times10^4$ | | Burkholder et al. (2019) | L | |
| $C_4H_{10}O_2$ | $3.5\times10^4$ | | Burkholder et al. (2015) | L | |
| [110-63-4] | $>9.0\times10^2$ | | Altschuh et al. (1999) | M | |
| WERYXYBDKMZEQL-UHFFFAOYSA-N | $7.6\times10^3$ | | Duchowicz et al. (2020) | V | 186 |
| | $7.6\times10^3$ | | HSDB (2015) | V | |
| | $3.5\times10^4$ | 11000 | Compernolle and Müller (2014b) | V | |
| | $2.4\times10^3$ | | Duchowicz et al. (2020) | Q | |
| | $6.0\times10^2$ | | Wang et al. (2017) | Q | 80, 238 |
| | $7.6\times10^3$ | | Wang et al. (2017) | Q | 80, 239 |
| | $9.6\times10^3$ | | Wang et al. (2017) | Q | 80, 240 |
| | $3.1\times10^4$ | | Raventos-Duran et al. (2010) | Q | 242, 243 |
| | $4.9\times10^3$ | | Raventos-Duran et al. (2010) | Q | 244 |
| | $3.9\times10^1$ | | Raventos-Duran et al. (2010) | Q | 245 |
| | $8.0\times10^3$ | | Hilal et al. (2008) | Q | |
| | $1.6\times10^3$ | | Modarresi et al. (2007) | Q | 67 |
| | | | Saxena and Hildemann (1996) | E | 401, 432 |
| | $1.9\times10^4$ | | Yaws (1999) | ? | 21, 12 |
| 2,3-butanediol | $1.1\times10^3$ | | Burkholder et al. (2019) | L | |
| $C_4H_{10}O_2$ | $1.1\times10^3$ | | Burkholder et al. (2015) | L | |
| [513-85-9] | $3.4\times10^2$ | | Duchowicz et al. (2020) | V | 186 |
| OWBTYPJTUOEWEK-UHFFFAOYSA-N | $3.4\times10^2$ | | HSDB (2015) | V | |
| | $1.1\times10^3$ | 9900 | Compernolle and Müller (2014b) | V | |
| | $9.7\times10^2$ | | Yaws (2003) | X | 237, 12 |
| | $1.7\times10^2$ | | Duchowicz et al. (2020) | Q | |
| | $3.3\times10^2$ | | Wang et al. (2017) | Q | 80, 238 |
| | $1.2\times10^3$ | | Wang et al. (2017) | Q | 80, 239 |
| | $4.1\times10^2$ | | Wang et al. (2017) | Q | 80, 240 |
| | $1.7\times10^3$ | | Gharagheizi et al. (2012) | Q | |
| | $7.8\times10^2$ | | Gharagheizi et al. (2010) | Q | 246 |
| | | | Saxena and Hildemann (1996) | E | 401, 433 |
| *meso*-2,3-butanediol | $2.2\times10^2$ | | Duchowicz et al. (2020) | V | 186 |
| $C_4H_{10}O_2$ | $6.8\times10^2$ | | Yaws (2003) | X | 237, 12 |
| [5341-95-7] | $1.7\times10^2$ | | Duchowicz et al. (2020) | Q | |
| OWBTYPJTUOEWEK-ZXZARUISSA-N | $1.6\times10^3$ | | Gharagheizi et al. (2012) | Q | |
| | $7.8\times10^2$ | | Gharagheizi et al. (2010) | Q | 246 |



Table A3.3: Polyols $(R(OH)_n)$ (... continued)

| Substance Formula (Trivial Name) [CAS Registry Number] InChIKey | $H_s^{cp}$ (at $T^\ominus$) $\left[\dfrac{\text{mol}}{\text{m}^3\,\text{Pa}}\right]$ | $\dfrac{\text{d}\ln H_s^{cp}}{\text{d}(1/T)}$ [K] | Reference | Type | Note |
|---|---|---|---|---|---|
| 2-methylpropane-1,2-diol $C_4H_{10}O_2$ [558-43-0] BTVWZWFKMIUSGS-UHFFFAOYSA-N | $1.8\times10^2$ $5.1\times10^2$ $3.2\times10^2$ | | Wang et al. (2017) Wang et al. (2017) Wang et al. (2017) | Q Q Q | 80, 238 80, 239 80, 240 |
| 2-methylpropane-1,3-diol $C_4H_{10}O_2$ [2163-42-0] QWGRWMMWNDWRQN-UHFFFAOYSA-N | $6.9\times10^2$ $5.0\times10^3$ $2.6\times10^3$ $4.3\times10^1$ | | Wang et al. (2017) Wang et al. (2017) Wang et al. (2017) HSDB (2015) | Q Q Q Q | 80, 238 80, 239 80, 240 99 |
| 1,2,3-butanetriol $C_4H_{10}O_3$ [4435-50-1] YAXKTBLXMTYWDQ-UHFFFAOYSA-N | $3.0\times10^9$ | | Saxena and Hildemann (1996) | E | 401 |
| 1,2,4-butanetriol $C_4H_{10}O_3$ [3068-00-6] ARXKVVRQIIOZGF-UHFFFAOYSA-N | $7.1\times10^5$ $1.9\times10^6$ $3.2\times10^5$ $3.0\times10^9$ | | Wang et al. (2017) Wang et al. (2017) Wang et al. (2017) Saxena and Hildemann (1996) | Q Q Q E | 80, 238 80, 239 80, 240 401 |
| 1,2,3,4-butanetetrol $C_4H_{10}O_4$ (1,2,3,4-tetrahydroxybutane; tetritol) [7541-59-5] UNXHWFMMPAWVPI-UHFFFAOYSA-N | $2.0\times10^{14}$ | | Saxena and Hildemann (1996) | E | 401 |
| $2(R),3(S)$-1,2,3,4-butanetetrol $C_4H_{10}O_4$ (erythritol) [149-32-6] UNXHWFMMPAWVPI-ZXZARUISSA-N | $1.1\times10^{10}$ $1.1\times10^{10}$ $2.5\times10^9$ $1.1\times10^{10}$ $3.2\times10^4$ $4.1\times10^8$ $3.2\times10^4$ | 16000 | Burkholder et al. (2019) Burkholder et al. (2015) Qin et al. (2021) Compernolle and Müller (2014b) Qin et al. (2021) Qin et al. (2021) HSDB (2015) | L L M V Q Q Q | 434 435 436 99 |
| 1,2-pentanediol $C_5H_{12}O_2$ [5343-92-0] WCVRQHFDJLLWFE-UHFFFAOYSA-N | $1.4\times10^3$ $2.2\times10^2$ $5.9\times10^2$ $4.2\times10^2$ | | Compernolle and Müller (2014b) Wang et al. (2017) Wang et al. (2017) Wang et al. (2017) | V Q Q Q | 80, 238 80, 239 80, 240 |
| 1,3-pentanediol $C_5H_{12}O_2$ [3174-67-2] RUOPINZRYMFPBF-UHFFFAOYSA-N | $5.6\times10^2$ $3.1\times10^3$ $1.5\times10^3$ | | Wang et al. (2017) Wang et al. (2017) Wang et al. (2017) | Q Q Q | 80, 238 80, 239 80, 240 |
| 1,4-pentanediol $C_5H_{12}O_2$ [626-95-9] GLOBUAZSRIOKLN-UHFFFAOYSA-N | $2.3\times10^4$ $5.6\times10^2$ $3.9\times10^3$ $4.8\times10^3$ | | Compernolle and Müller (2014b) Wang et al. (2017) Wang et al. (2017) Wang et al. (2017) | V Q Q Q | 80, 238 80, 239 80, 240 |



Table A3.3: Polyols $(R(OH)_n)$ (... continued)

| Substance Formula (Trivial Name) [CAS Registry Number] InChIKey | $H_s^{cp}$ (at $T^{\ominus}$) $\left[\dfrac{\mathrm{mol}}{\mathrm{m^3\,Pa}}\right]$ | $\dfrac{\mathrm{d}\ln H_s^{cp}}{\mathrm{d}(1/T)}$ [K] | Reference | Type | Note |
|---|---|---|---|---|---|
| 1,5-pentanediol $C_5H_{12}O_2$ [111-29-5] ALQSHHUCVQOPAS-UHFFFAOYSA-N | $7.0\times10^4$ $2.0\times10^4$ $1.6\times10^4$ $3.1\times10^1$ $7.7\times10^3$ $3.9\times10^4$ | 12000 | Compernolle and Müller (2014b) Raventos-Duran et al. (2010) Raventos-Duran et al. (2010) Raventos-Duran et al. (2010) Hilal et al. (2008) Saxena and Hildemann (1996) | V Q Q Q Q E | 242, 243 244 245 401 |
| 2,3-pentanediol $C_5H_{12}O_2$ [42027-23-6] XLMFDCKSFJWJTP-UHFFFAOYSA-N | $2.6\times10^2$ $1.2\times10^3$ $1.7\times10^2$ $3.0\times10^4$ | | Wang et al. (2017) Wang et al. (2017) Wang et al. (2017) Saxena and Hildemann (1996) | Q Q Q E | 80, 238 80, 239 80, 240 401 |
| 2,4-pentanediol $C_5H_{12}O_2$ [625-69-4] GTCCGKPBSJZVRZ-UHFFFAOYSA-N | $3.8\times10^3$ $6.5\times10^2$ $5.4\times10^3$ $2.3\times10^3$ $3.0\times10^4$ | | Compernolle and Müller (2014b) Wang et al. (2017) Wang et al. (2017) Wang et al. (2017) Saxena and Hildemann (1996) | V Q Q Q E | 80, 238 80, 239 80, 240 401 |
| 3-methylbutane-1,2-diol $C_5H_{12}O_2$ [50468-22-9] HJJZIMFAIMUSBW-UHFFFAOYSA-N | $2.6\times10^2$ $8.5\times10^2$ $3.3\times10^2$ | | Wang et al. (2017) Wang et al. (2017) Wang et al. (2017) | Q Q Q | 80, 238 80, 239 80, 240 |
| 3-methylbutane-1,3-diol $C_5H_{12}O_2$ [2568-33-4] XPFCZYUVICHKDS-UHFFFAOYSA-N | $4.0\times10^2$ $2.6\times10^3$ $2.1\times10^3$ | | Wang et al. (2017) Wang et al. (2017) Wang et al. (2017) | Q Q Q | 80, 238 80, 239 80, 240 |
| 2-methylbutane-1,2-diol $C_5H_{12}O_2$ [41051-72-3] DOPZLYNWNJHAOS-UHFFFAOYSA-N | $1.6\times10^2$ $4.3\times10^2$ $1.1\times10^2$ | | Wang et al. (2017) Wang et al. (2017) Wang et al. (2017) | Q Q Q | 80, 238 80, 239 80, 240 |
| 2-methylbutane-1,3-diol $C_5H_{12}O_2$ [684-84-4] GNBPEYCZELNJMS-UHFFFAOYSA-N | $6.5\times10^2$ $4.7\times10^3$ $2.0\times10^3$ | | Wang et al. (2017) Wang et al. (2017) Wang et al. (2017) | Q Q Q | 80, 238 80, 239 80, 240 |
| 2-methylbutane-1,4-diol $C_5H_{12}O_2$ [2938-98-9] MWCBGWLCXSUTHK-UHFFFAOYSA-N | $5.6\times10^2$ $6.0\times10^3$ $6.2\times10^3$ | | Wang et al. (2017) Wang et al. (2017) Wang et al. (2017) | Q Q Q | 80, 238 80, 239 80, 240 |
| 2-methylbutane-2,3-diol $C_5H_{12}O_2$ [5396-58-7] IDEOPBXRUBNYBN-UHFFFAOYSA-N | $1.8\times10^2$ $7.3\times10^2$ $2.0\times10^2$ | | Wang et al. (2017) Wang et al. (2017) Wang et al. (2017) | Q Q Q | 80, 238 80, 239 80, 240 |



Table A3.3: Polyols ($R(OH)_n$) (...continued)

| Substance Formula (Trivial Name) [CAS Registry Number] InChIKey | $H_s^{cp}$ (at $T^\ominus$) $\left[\dfrac{\mathrm{mol}}{\mathrm{m}^3\,\mathrm{Pa}}\right]$ | $\dfrac{\mathrm{d}\ln H_s^{cp}}{\mathrm{d}(1/T)}$ [K] | Reference | Type | Note |
|---|---|---|---|---|---|
| 2,2-dimethylpropane-1,3-diol $C_5H_{12}O_2$ [126-30-7] SLCVBVWXLSEKPL-UHFFFAOYSA-N | $4.0\times10^2$ $3.1\times10^3$ $6.8\times10^2$ | | Wang et al. (2017) Wang et al. (2017) Wang et al. (2017) | Q Q Q | 80, 238 80, 239 80, 240 |
| 2-(hydroxymethyl)-2-methyl-1,3-propanediol $C_5H_{12}O_3$ [77-85-0] QXJQHYBHAIHNGG-UHFFFAOYSA-N | $9.0\times10^2$ | | HSDB (2015) | Q | 99 |
| 2-methylbutane-1,2,4-triol $C_5H_{12}O_3$ [62875-07-4] XYHGSPUTABMVOC-UHFFFAOYSA-N | $3.9\times10^5$ $1.0\times10^6$ $1.7\times10^5$ | | Wang et al. (2017) Wang et al. (2017) Wang et al. (2017) | Q Q Q | 80, 238 80, 239 80, 240 |
| MCM:C56OH $C_5H_{12}O_3$ RXEJCNRKXVSXDJ-UHFFFAOYSA-N | $6.6\times10^5$ $2.0\times10^6$ $8.1\times10^4$ | | Wang et al. (2017) Wang et al. (2017) Wang et al. (2017) | Q Q Q | 80, 238 80, 239 80, 240 |
| MCM:HO124C5 $C_5H_{12}O_3$ MOIOWCZVZKHQIC-UHFFFAOYSA-N | $6.6\times10^5$ $1.3\times10^6$ $1.5\times10^5$ | | Wang et al. (2017) Wang et al. (2017) Wang et al. (2017) | Q Q Q | 80, 238 80, 239 80, 240 |
| MCM:HO134C5 $C_5H_{12}O_3$ ANUUQAHHEZMTAS-UHFFFAOYSA-N | $6.6\times10^5$ $3.2\times10^6$ $1.4\times10^5$ | | Wang et al. (2017) Wang et al. (2017) Wang et al. (2017) | Q Q Q | 80, 238 80, 239 80, 240 |
| MCM:MBOAOH $C_5H_{12}O_3$ QFZITDCVRJQLMZ-UHFFFAOYSA-N | $3.0\times10^5$ $1.1\times10^6$ $5.4\times10^3$ | | Wang et al. (2017) Wang et al. (2017) Wang et al. (2017) | Q Q Q | 80, 238 80, 239 80, 240 |
| 2-pentene-2,4,5-triol $C_5H_{10}O_3$ (ME1TRIOL) MTMASQITBHTGFW-UHFFFAOYSA-N | $4.8\times10^7$ | 11000 | Wieser et al. (2023) | Q | 437 |
| 2-methyl-3-butene-1,2,3-triol $C_5H_{10}O_3$ YXPJRFSYSVRDQI-UHFFFAOYSA-N | $3.9\times10^8$ $1.5\times10^2$ $3.1\times10^2$ | | Qin et al. (2021) Qin et al. (2021) Qin et al. (2021) | M Q Q | 434 435 436 |
| (E)-2-methyl-1-butene-1,3,4-triol $C_5H_{10}O_3$ FVYHKVMWBMOFGS-DUXPYHPUSA-N | $6.2\times10^8$ $1.3\times10^2$ $3.7\times10^6$ | | Qin et al. (2021) Qin et al. (2021) Qin et al. (2021) | M Q Q | 434 435 436 |
| (Z)-2-methyl-1-butene-1,3,4-triol $C_5H_{10}O_3$ FVYHKVMWBMOFGS-RQOWECAXSA-N | $7.8\times10^8$ $1.3\times10^2$ $3.7\times10^6$ | | Qin et al. (2021) Qin et al. (2021) Qin et al. (2021) | M Q Q | 434 435 436 |
| 2-methylthreitol $C_5H_{12}O_4$ HGVJFBSSLICXEM-WHFBIAKZSA-N | $2.0\times10^8$ $2.4\times10^4$ $3.3\times10^8$ | | Qin et al. (2021) Qin et al. (2021) Qin et al. (2021) | M Q Q | 434 435 436 |





Table A3.3: Polyols ($R(OH)_n$) (...continued)

| Substance<br>Formula<br>(Trivial Name)<br>[CAS Registry Number]<br>InChIKey | $H_s^{cp}$<br>(at $T^{\ominus}$)<br>$\left[\dfrac{\mathrm{mol}}{\mathrm{m^3\,Pa}}\right]$ | $\dfrac{\mathrm{d}\ln H_s^{cp}}{\mathrm{d}(1/T)}$<br><br>[K] | Reference | Type | Note |
|---|---|---|---|---|---|
| 2-methylerythritol<br>$C_5H_{12}O_4$<br>[93921-83-6]<br>HGVJFBSSLICXEM-CRCLSJGQSA-N | $3.1\times10^8$<br>$2.4\times10^4$<br>$3.3\times10^8$<br>$6.2\times10^{10}$<br>$3.1\times10^8$<br>$3.1\times10^4$ | | Qin et al. (2021)<br>Qin et al. (2021)<br>Qin et al. (2021)<br>Isaacman-VanWertz et al. (2016)<br>Isaacman-VanWertz et al. (2016)<br>Isaacman-VanWertz et al. (2016) | M<br>Q<br>Q<br>Q<br>Q<br>Q | 434<br>435<br>436<br>438, 439<br>440, 441<br>442 |
| 2-Methylbutane-1,2,3,4-tetrol<br>$C_5H_{12}O_4$<br>(MeBuTETROL)<br>[42933-13-1]<br>HGVJFBSSLICXEM-UHFFFAOYSA-N | $6.1\times10^{10}$ | 14000 | Wieser et al. (2023) | Q | 437 |
| 2,2-bis(hydroxymethyl)1,3-propanediol<br>$C_5H_{12}O_4$<br>(pentaerythritol)<br>[115-77-5]<br>WXZMFSXDPGVJKK-UHFFFAOYSA-N | $7.3\times10^{10}$<br><br>$7.3\times10^{10}$<br>$7.3\times10^{10}$<br>$1.5\times10^9$<br>$2.4\times10^4$<br>$1.2\times10^8$<br>$2.4\times10^9$ | <br><br><br>16000 | Burkholder et al. (2019)<br><br>Burkholder et al. (2015)<br>Compernolle and Müller (2014b)<br>Yaws (2003)<br>HSDB (2015)<br>Gharagheizi et al. (2012)<br>Gharagheizi et al. (2010) | L<br><br>L<br>V<br>X<br>Q<br>Q<br>Q | <br><br><br><br>237, 12<br>99<br><br>246 |
| 1,2,3,4,5-pentanepentol<br>$C_5H_{12}O_5$<br>[7643-75-6]<br>HEBKCHPVOIAQTA-UHFFFAOYSA-N | $8.9\times10^{18}$ | | Saxena and Hildemann (1996) | E | 401 |
| (2R,3R,4S)-pentane-1,2,3,4,5-pentol<br>$C_5H_{12}O_5$<br>(xylitol)<br>[87-99-0]<br>HEBKCHPVOIAQTA-SCDXWVJYSA-N | $4.9\times10^8$<br><br>$3.9\times10^{11}$<br>$6.6\times10^5$<br>$7.3\times10^{11}$<br>$6.6\times10^5$ | <br><br>17000 | Qin et al. (2021)<br><br>Compernolle and Müller (2014b)<br>Qin et al. (2021)<br>Qin et al. (2021)<br>HSDB (2015) | M<br><br>V<br>Q<br>Q<br>Q | 434<br><br><br>435<br>436<br>99 |
| (2R,3S,4S)-pentane-1,2,3,4,5-pentol<br>$C_5H_{12}O_5$<br>(adonitol; ribitol)<br>[488-81-3]<br>HEBKCHPVOIAQTA-ZXFHETKHSA-N | $4.6\times10^{11}$ | 18000 | Compernolle and Müller (2014b) | V | |
| (2R,4R)-pentane-1,2,3,4,5-pentol<br>$C_5H_{12}O_5$<br>(arabitol; arabinitol)<br>[2152-56-9]<br>HEBKCHPVOIAQTA-QWWZWVQMSA-N | $9.9\times10^9$<br>$6.7\times10^{11}$<br>$6.6\times10^5$<br>$7.3\times10^{11}$ | <br>18000 | Qin et al. (2021)<br>Compernolle and Müller (2014b)<br>Qin et al. (2021)<br>Qin et al. (2021) | M<br>V<br>Q<br>Q | 434<br><br>435<br>436 |



Table A3.3: Polyols ($R(OH)_n$) (... continued)

| Substance<br>Formula<br>(Trivial Name)<br>[CAS Registry Number]<br>InChIKey | $H_s^{cp}$<br>(at $T^\ominus$)<br>$\left[\dfrac{\text{mol}}{\text{m}^3\,\text{Pa}}\right]$ | $\dfrac{\mathrm{d}\ln H_s^{cp}}{\mathrm{d}(1/T)}$<br><br>[K] | Reference | Type | Note |
|---|---|---|---|---|---|
| levoglucosan<br>$C_6H_{10}O_5$<br>[498-07-7]<br>TWNIBLMWSKIRAT-VFUOTHLCSA-N | $4.9\times10^{10}$<br>$6.9\times10^{7}$<br>$4.3\times10^{10}$ | | Qin et al. (2021)<br>Qin et al. (2021)<br>Qin et al. (2021) | M<br>Q<br>Q | 434<br>435<br>436 |
| fructose<br>$C_6H_{12}O_6$<br>[30237-26-4]<br>BJHIKXHVCXFQLS-UYFOZJQFSA-N | $1.2\times10^{9}$<br>$1.0\times10^{9}$<br>$1.6\times10^{14}$ | | Qin et al. (2021)<br>Qin et al. (2021)<br>Qin et al. (2021) | M<br>Q<br>Q | 434<br>435<br>436 |
| mannose<br>$C_6H_{12}O_6$<br>[3458-28-4]<br>GZCGUPFRVQAUEE-KVTDHHQDSA-N | $1.2\times10^{9}$<br>$1.7\times10^{5}$<br>$1.4\times10^{13}$ | | Qin et al. (2021)<br>Qin et al. (2021)<br>Qin et al. (2021) | M<br>Q<br>Q | 434<br>435<br>436 |
| glucose<br>$C_6H_{12}O_6$<br>[50-99-7]<br>GZCGUPFRVQAUEE-SLPGGIOYSA-N | $7.8\times10^{8}$<br>$1.0\times10^{9}$<br>$1.8\times10^{15}$ | | Qin et al. (2021)<br>Qin et al. (2021)<br>Qin et al. (2021) | M<br>Q<br>Q | 434<br>435<br>436 |
| 1,2-hexanediol<br>$C_6H_{14}O_2$<br>[6920-22-5]<br>FHKSXSQHXQEMOK-UHFFFAOYSA-N | $1.7\times10^{3}$<br>$1.9\times10^{2}$<br>$4.4\times10^{2}$<br>$4.2\times10^{2}$ | | Compernolle and Müller (2014b)<br>Wang et al. (2017)<br>Wang et al. (2017)<br>Wang et al. (2017) | V<br>Q<br>Q<br>Q | <br>80, 238<br>80, 239<br>80, 240 |
| 1,4-hexanediol<br>$C_6H_{14}O_2$<br>[16432-53-4]<br>QVTWBMUAJHVAIJ-UHFFFAOYSA-N | $4.5\times10^{2}$<br>$3.3\times10^{3}$<br>$1.6\times10^{3}$ | | Wang et al. (2017)<br>Wang et al. (2017)<br>Wang et al. (2017) | Q<br>Q<br>Q | 80, 238<br>80, 239<br>80, 240 |
| 1,6-hexanediol<br>$C_6H_{14}O_2$<br>[629-11-8]<br>XXMIOPMDWAUFGU-UHFFFAOYSA-N | $4.5\times10^{4}$<br>$3.0\times10^{4}$ | | HSDB (2015)<br>Saxena and Hildemann (1996) | Q<br>E | 99<br>401 |
| 2,3-hexanediol<br>$C_6H_{14}O_2$<br>[617-30-1]<br>QCIYAEYRVFUFAP-UHFFFAOYSA-N | $2.1\times10^{2}$<br>$8.7\times10^{2}$<br>$1.3\times10^{2}$ | | Wang et al. (2017)<br>Wang et al. (2017)<br>Wang et al. (2017) | Q<br>Q<br>Q | 80, 238<br>80, 239<br>80, 240 |
| 2,5-hexanediol<br>$C_6H_{14}O_2$<br>[2935-44-6]<br>OHMBHFSEKCCCBW-UHFFFAOYSA-N | $1.4\times10^{4}$<br>$5.1\times10^{2}$<br>$5.9\times10^{3}$<br>$2.0\times10^{3}$<br>$2.0\times10^{4}$ | | Compernolle and Müller (2014b)<br>Wang et al. (2017)<br>Wang et al. (2017)<br>Wang et al. (2017)<br>Saxena and Hildemann (1996) | V<br>Q<br>Q<br>Q<br>E | <br>80, 238<br>80, 239<br>80, 240<br>401 |
| 1,2-cyclohexanediol<br>$C_6H_{12}O_2$<br>[931-17-9]<br>PFURGBBHAOXLIO-UHFFFAOYSA-N | $6.6\times10^{2}$<br>$1.4\times10^{4}$<br>$1.3\times10^{3}$ | | Wang et al. (2017)<br>Wang et al. (2017)<br>Wang et al. (2017) | Q<br>Q<br>Q | 80, 238<br>80, 239<br>80, 240 |





Table A3.3: Polyols $(R(OH)_n)$ (...continued)

| Substance Formula (Trivial Name) [CAS Registry Number] InChIKey | $H_s^{cp}$ (at $T^\ominus$) $\left[\dfrac{\text{mol}}{\text{m}^3\,\text{Pa}}\right]$ | $\dfrac{\text{d}\ln H_s^{cp}}{\text{d}(1/T)}$ [K] | Reference | Type | Note |
|---|---|---|---|---|---|
| 2-methyl-1,3-pentanediol $C_6H_{14}O_2$ [149-31-5] SPXWGAHNKXLXAP-UHFFFAOYSA-N | $3.0\times10^4$ | | Saxena and Hildemann (1996) | E | 401 |
| 2-methyl-1,4-pentanediol $C_6H_{14}O_2$ [6287-17-8] PNJNLCNHYSWUPT-UHFFFAOYSA-N | $5.1\times10^2$ $4.0\times10^3$ $4.1\times10^3$ | | Wang et al. (2017) Wang et al. (2017) Wang et al. (2017) | Q Q Q | 80, 238 80, 239 80, 240 |
| 2-methyl-2,4-pentanediol $C_6H_{14}O_2$ [107-41-5] SVTBMSDMJJWYQN-UHFFFAOYSA-N | $2.5\times10^1$ $2.0\times10^4$ | | HSDB (2015) Saxena and Hildemann (1996) | Q E | 99 401 |
| 3-methyl-1,3-pentanediol $C_6H_{14}O_2$ [33879-72-0] HIYKOZFIVZIBFO-UHFFFAOYSA-N | $3.1\times10^2$ $2.3\times10^3$ $8.9\times10^2$ | | Wang et al. (2017) Wang et al. (2017) Wang et al. (2017) | Q Q Q | 80, 238 80, 239 80, 240 |
| 3-methyl-1,4-pentanediol $C_6H_{14}O_2$ [26787-63-3] WOHXXIWTEHLCQK-UHFFFAOYSA-N | $5.1\times10^2$ $5.8\times10^3$ $2.6\times10^3$ | | Wang et al. (2017) Wang et al. (2017) Wang et al. (2017) | Q Q Q | 80, 238 80, 239 80, 240 |
| 4-methyl-1,4-pentanediol $C_6H_{14}O_2$ [1462-10-8] HAIVWDGLCRYQMC-UHFFFAOYSA-N | $3.1\times10^2$ $2.9\times10^3$ $2.5\times10^3$ | | Wang et al. (2017) Wang et al. (2017) Wang et al. (2017) | Q Q Q | 80, 238 80, 239 80, 240 |
| 2,3-dimethyl-2,3-butanediol $C_6H_{14}O_2$ [76-09-5] IVDFJHOHABJVEH-UHFFFAOYSA-N | $1.0\times10^2$ $8.7\times10^2$ $6.9\times10^1$ | | Wang et al. (2017) Wang et al. (2017) Wang et al. (2017) | Q Q Q | 80, 238 80, 239 80, 240 |
| 2,3-dimethyl-1,4-butanediol $C_6H_{14}O_2$ [57716-80-0] SKQUTIPQJKQFRA-UHFFFAOYSA-N | $5.1\times10^2$ $8.1\times10^3$ $2.8\times10^3$ | | Wang et al. (2017) Wang et al. (2017) Wang et al. (2017) | Q Q Q | 80, 238 80, 239 80, 240 |
| 2,2-dimethyl-1,4-butanediol $C_6H_{14}O_2$ [32812-23-0] GQSZUUPRBBBHRI-UHFFFAOYSA-N | $3.1\times10^2$ $4.5\times10^3$ $2.5\times10^3$ | | Wang et al. (2017) Wang et al. (2017) Wang et al. (2017) | Q Q Q | 80, 238 80, 239 80, 240 |
| 1,2,6-hexanetriol $C_6H_{14}O_3$ [106-69-4] ZWVMLYRJXORSEP-UHFFFAOYSA-N | $2.0\times10^9$ | | Saxena and Hildemann (1996) | E | 401 |





Table A3.3: Polyols ($R(OH)_n$) (...continued)

| Substance<br>Formula<br>(Trivial Name)<br>[CAS Registry Number]<br>InChIKey | $H_s^{cp}$<br>(at $T^\ominus$)<br><br>$\left[\dfrac{\text{mol}}{\text{m}^3\,\text{Pa}}\right]$ | $\dfrac{\text{d}\ln H_s^{cp}}{\text{d}(1/T)}$<br><br>[K] | Reference | Type | Note |
|---|---|---|---|---|---|
| MCM:H134M3C5<br>$C_6H_{14}O_3$<br>XPUHQLFLUQJINS-UHFFFAOYSA-N | $3.6\times10^5$<br>$2.4\times10^6$<br>$8.3\times10^4$ | | Wang et al. (2017)<br>Wang et al. (2017)<br>Wang et al. (2017) | Q<br>Q<br>Q | 80, 238<br>80, 239<br>80, 240 |
| 1,2,3,4,5,6-hexahydroxy hexane<br>$C_6H_{14}O_6$<br>[45007-61-2]<br>FBPFZTCFMRRESA-UHFFFAOYSA-N | $3.9\times10^{23}$ | | Saxena and Hildemann (1996) | E | 401 |
| (2S,3R,4R,5R)-hexane-1,2,3,4,5,6-hexol<br>$C_6H_{14}O_6$<br>(sorbitol)<br>[50-70-4]<br>FBPFZTCFMRRESA-JGWLITMVSA-N | $6.6\times10^{14}$<br><br>$1.4\times10^7$ | 22000 | Compernolle and Müller (2014b)<br><br>HSDB (2015) | V<br><br>Q | <br><br>99 |
| (2R,3R,4R,5R)-hexane-1,2,3,4,5,6-hexol<br>$C_6H_{14}O_6$<br>(mannitol)<br>[69-65-8]<br>FBPFZTCFMRRESA-KVTDHHQDSA-N | $3.1\times10^9$<br>$1.8\times10^{15}$<br>$1.4\times10^7$<br>$1.0\times10^{15}$<br>$1.4\times10^7$ | 22000 | Qin et al. (2021)<br>Compernolle and Müller (2014b)<br>Qin et al. (2021)<br>Qin et al. (2021)<br>HSDB (2015) | M<br>V<br>Q<br>Q<br>Q | 434<br><br>435<br>436<br>99 |
| (2R,3S,4R,5S)-hexane-1,2,3,4,5,6-hexol<br>$C_6H_{14}O_6$<br>(dulcitol; galactitol)<br>[608-66-2]<br>FBPFZTCFMRRESA-GUCUJZIJSA-N | $9.0\times10^{14}$ | 22000 | Compernolle and Müller (2014b) | V | |
| 1,2,4,5-cyclohexanetetrol<br>$C_6H_{12}O_4$<br>(1,2,4,5-tetrahydroxycyclohexane)<br>[35652-37-0]<br>RDIDGZFQASQXBU-UHFFFAOYSA-N | $3.9\times10^{14}$ | | Saxena and Hildemann (1996) | E | 401 |
| 1,2,3,4,5,6-hexahydroxycyclohexane<br>$C_6H_{12}O_6$<br>[87-89-8]<br>CDAISMWEOUEBRE-UHFFFAOYSA-N | $9.9\times10^{23}$ | | Saxena and Hildemann (1996) | E | 401 |
| 1,7-heptanediol<br>$C_7H_{16}O_2$<br>[629-30-1]<br>SXCBDZAEHILGLM-UHFFFAOYSA-N | <br>$2.0\times10^4$ | | Compernolle and Müller (2014b)<br>Saxena and Hildemann (1996) | V<br>E | 443<br>401 |
| 2,4-heptanediol<br>$C_7H_{16}O_2$<br>[20748-86-1]<br>XVEOUOTUJBYHNL-UHFFFAOYSA-N | $2.0\times10^4$ | | Saxena and Hildemann (1996) | E | 401 |



Table A3.3: Polyols $(R(OH)_n)$ (...continued)

| Substance Formula (Trivial Name) [CAS Registry Number] InChIKey | $H_s^{cp}$ (at $T^{\ominus}$) $\left[\dfrac{\text{mol}}{\text{m}^3\,\text{Pa}}\right]$ | $\dfrac{\text{d}\ln H_s^{cp}}{\text{d}(1/T)}$ [K] | Reference | Type | Note |
|---|---|---|---|---|---|
| 2,5-heptanediol $C_7H_{16}O_2$ [70444-25-6] XTVHTJKQKUOEQA-UHFFFAOYSA-N | $4.2\times10^2$ $4.7\times10^3$ $7.3\times10^2$ | | Wang et al. (2017) Wang et al. (2017) Wang et al. (2017) | Q Q Q | 80, 238 80, 239 80, 240 |
| 4-methyl-1,4-hexanediol $C_7H_{16}O_2$ [40646-08-0] ZRCYHJNCCBSTSZ-UHFFFAOYSA-N | $2.5\times10^2$ $2.6\times10^3$ $1.2\times10^3$ | | Wang et al. (2017) Wang et al. (2017) Wang et al. (2017) | Q Q Q | 80, 238 80, 239 80, 240 |
| 2-methyl-2,5-hexanediol $C_7H_{16}O_2$ [29044-06-2] KZWQVDXGMOSSNY-UHFFFAOYSA-N | $2.9\times10^2$ $4.1\times10^3$ $1.2\times10^3$ | | Wang et al. (2017) Wang et al. (2017) Wang et al. (2017) | Q Q Q | 80, 238 80, 239 80, 240 |
| 3-methyl-2,5-hexanediol $C_7H_{16}O_2$ PMORVUNYTOZCSE-UHFFFAOYSA-N | $4.8\times10^2$ $7.6\times10^3$ $2.2\times10^3$ | | Wang et al. (2017) Wang et al. (2017) Wang et al. (2017) | Q Q Q | 80, 238 80, 239 80, 240 |
| 2,2-diethyl-1,3-propanediol $C_7H_{16}O_2$ [115-76-4] XRVCFZPJAHWYTB-UHFFFAOYSA-N | $2.0\times10^4$ | | Saxena and Hildemann (1996) | E | 401 |
| 1,2,3,4,5-pentahydroxyheptane $C_7H_{16}O_5$ HUYORHVLGRMTGF-UHFFFAOYSA-N | $4.9\times10^{18}$ | | Saxena and Hildemann (1996) | E | 401 |
| 1,2,3,4,6-pentahydroxyheptane $C_7H_{16}O_5$ HBKKLPCMKBGRJH-UHFFFAOYSA-N | $3.9\times10^{18}$ | | Saxena and Hildemann (1996) | E | 401 |
| 1,2,3,5,7-pentahydroxyheptane $C_7H_{16}O_5$ NSOOVWLSFZLKQX-UHFFFAOYSA-N | $4.9\times10^{18}$ | | Saxena and Hildemann (1996) | E | 401 |
| 1,2,3,4,5,6-hexahydroxyheptane $C_7H_{16}O_6$ (1-deoxy-heptitol) [688007-16-1] YMEXGEAJNZRQEH-VIFPVBQESA-N | $3.0\times10^{23}$ | | Saxena and Hildemann (1996) | E | 401 |
| 3,6-octanediol $C_8H_{18}O_2$ [24434-09-1] BCKOQWWRTRBSGR-UHFFFAOYSA-N | $3.7\times10^2$ $4.1\times10^3$ $2.1\times10^2$ | | Wang et al. (2017) Wang et al. (2017) Wang et al. (2017) | Q Q Q | 80, 238 80, 239 80, 240 |
| 1,4-cyclohexanedimethanol $C_8H_{16}O_2$ [105-08-8] YIMQCDZDWXUDCA-UHFFFAOYSA-N | $1.5\times10^5$ | | HSDB (2015) | V | |



Table A3.3: Polyols $(R(OH)_n)$ (...continued)

| Substance Formula (Trivial Name) [CAS Registry Number] InChIKey | $H_s^{cp}$ (at $T^{\ominus}$) $\left[\dfrac{\mathrm{mol}}{\mathrm{m}^3\,\mathrm{Pa}}\right]$ | $\dfrac{\mathrm{d}\ln H_s^{cp}}{\mathrm{d}(1/T)}$ [K] | Reference | Type | Note |
|---|---|---|---|---|---|
| 2-ethyl-1,3-hexanediol $C_8H_{18}O_2$ [94-96-2] RWLALWYNXFYRGW-UHFFFAOYSA-N | $7.2 \times 10^2$ $5.4 \times 10^2$ $7.8 \times 10^2$ $7.8 \times 10^2$ $1.6 \times 10^1$ $1.1 \times 10^2$ $7.5 \times 10^2$ $2.0 \times 10^4$ | | Duchowicz et al. (2020) Duchowicz et al. (2020) Raventos-Duran et al. (2010) Raventos-Duran et al. (2010) Raventos-Duran et al. (2010) Hilal et al. (2008) Modarresi et al. (2007) Saxena and Hildemann (1996) | V Q Q Q Q Q Q E | 186  242, 243 244 245  67 401 |
| 2,2,4-trimethyl-1,3-pentanediol $C_8H_{18}O_2$ [144-19-4] JCTXKRPTIMZBJT-UHFFFAOYSA-N | $2.2 \times 10^2$ $9.2 \times 10^1$ $1.4 \times 10^1$ | | Duchowicz et al. (2020) Duchowicz et al. (2020) HSDB (2015) | V Q Q | 186  99 |
| 2,5-dimethyl-2,5-hexanediol $C_8H_{18}O_2$ [110-03-2] ZWNMRZQYWRLGMM-UHFFFAOYSA-N | $1.4 \times 10^1$ $1.9 \times 10^3$ $7.9 \times 10^2$ $4.7 \times 10^1$ | | Zhang et al. (2010) Zhang et al. (2010) Zhang et al. (2010) Zhang et al. (2010) | Q Q Q Q | 287, 288 287, 289 287, 290 287, 291 |
| 1,9-nonanediol $C_9H_{20}O_2$ [3937-56-2] ALVZNPYWJMLXKV-UHFFFAOYSA-N | | | Compernolle and Müller (2014b) | V | 444 |
| 3,6-nonanediol $C_9H_{20}O_2$ [4469-85-6] YKEWIFGQPMVBEF-UHFFFAOYSA-N | $3.0 \times 10^2$ $3.2 \times 10^3$ $1.8 \times 10^2$ | | Wang et al. (2017) Wang et al. (2017) Wang et al. (2017) | Q Q Q | 80, 238 80, 239 80, 240 |
| 1,10-decanediol $C_{10}H_{22}O_2$ [112-47-0] FOTKYAAJKYLFFN-UHFFFAOYSA-N | | | Compernolle and Müller (2014b) | V | 445 |
| 3,6-decanediol $C_{10}H_{22}O_2$ RBFBEPALGOYNGK-UHFFFAOYSA-N | $2.4 \times 10^2$ $2.6 \times 10^3$ $2.8 \times 10^2$ | | Wang et al. (2017) Wang et al. (2017) Wang et al. (2017) | Q Q Q | 80, 238 80, 239 80, 240 |
| MCM:APINBOH $C_{10}H_{18}O_2$ MOILFCKRQFQVFS-UHFFFAOYSA-N | $5.5 \times 10^2$ $3.2 \times 10^3$ $6.5 \times 10^3$ | | Wang et al. (2017) Wang et al. (2017) Wang et al. (2017) | Q Q Q | 80, 238 80, 239 80, 240 |
| MCM:APINCOH $C_{10}H_{18}O_2$ OMDMTHRBGUBUCO-UHFFFAOYSA-N | $1.8 \times 10^3$ $2.0 \times 10^5$ $3.0 \times 10^4$ | | Wang et al. (2017) Wang et al. (2017) Wang et al. (2017) | Q Q Q | 80, 238 80, 239 80, 240 |
| MCM:BPINAOH $C_{10}H_{18}O_2$ VXXAKGMRLUXFQH-UHFFFAOYSA-N | $5.5 \times 10^2$ $7.4 \times 10^2$ $1.0 \times 10^3$ | | Wang et al. (2017) Wang et al. (2017) Wang et al. (2017) | Q Q Q | 80, 238 80, 239 80, 240 |



Table A3.3: Polyols ($R(OH)_n$) (...continued)

| Substance Formula (Trivial Name) [CAS Registry Number] InChIKey | $H_s^{cp}$ (at $T^\ominus$) $\left[\dfrac{\text{mol}}{\text{m}^3\,\text{Pa}}\right]$ | $\dfrac{\text{d}\ln H_s^{cp}}{\text{d}(1/T)}$ [K] | Reference | Type | Note |
|---|---|---|---|---|---|
| MCM:BPINCOH $C_{10}H_{18}O_2$ XYKGEKWHBMLSGS-UHFFFAOYSA-N | $1.6\times10^3$ $3.5\times10^5$ $5.5\times10^4$ | | Wang et al. (2017) Wang et al. (2017) Wang et al. (2017) | Q Q Q | 80, 238 80, 239 80, 240 |
| MCM:HO36C11 $C_{11}H_{24}O_2$ VCLRIEYYOZNUNU-UHFFFAOYSA-N | $2.2\times10^2$ $2.0\times10^3$ $2.2\times10^2$ | | Wang et al. (2017) Wang et al. (2017) Wang et al. (2017) | Q Q Q | 80, 238 80, 239 80, 240 |
| MCM:HO36C12 $C_{12}H_{26}O_2$ CLQOYXKLYNGAHW-UHFFFAOYSA-N | $1.7\times10^2$ $1.6\times10^3$ $1.5\times10^2$ | | Wang et al. (2017) Wang et al. (2017) Wang et al. (2017) | Q Q Q | 80, 238 80, 239 80, 240 |
| 2-butene-1,4-diol $C_4H_8O_2$ [110-64-5] ORTVZLZNOYNASJ-UHFFFAOYSA-N | $>3.4\times10^2$ $2.5\times10^3$ $1.2\times10^5$ $3.4\times10^4$ | | Altschuh et al. (1999) Wang et al. (2017) Wang et al. (2017) Wang et al. (2017) | M Q Q Q | 80, 238 80, 239 80, 240 |
| 3-butene-1,2-diol $C_4H_8O_2$ [497-06-3] ITMIAZBRRZANGB-UHFFFAOYSA-N | $7.8\times10^2$ $8.5\times10^2$ $3.0\times10^2$ | | Wang et al. (2017) Wang et al. (2017) Wang et al. (2017) | Q Q Q | 80, 238 80, 239 80, 240 |
| 2-butyne-1,4-diol $C_4H_6O_2$ (1,4-dihydroxy-2-butyne) [110-65-6] DLDJFQGPPSQZKI-UHFFFAOYSA-N | $>2.0\times10^3$ $5.8\times10^5$ | | Altschuh et al. (1999) HSDB (2015) | M V | |
| MCM:C524OH $C_5H_{10}O_3$ PGARYUHLQUORKU-UHFFFAOYSA-N | $1.3\times10^6$ $6.5\times10^4$ $6.0\times10^4$ | | Wang et al. (2017) Wang et al. (2017) Wang et al. (2017) | Q Q Q | 80, 238 80, 239 80, 240 |
| MCM:ISOPAOH $C_5H_{10}O_2$ FLXLJBCLEUWWCG-UHFFFAOYSA-N | $1.6\times10^3$ $7.4\times10^4$ $3.4\times10^4$ | | Wang et al. (2017) Wang et al. (2017) Wang et al. (2017) | Q Q Q | 80, 238 80, 239 80, 240 |
| MCM:ISOPBOH $C_5H_{10}O_2$ XZRGYMKUQMPDQH-UHFFFAOYSA-N | $4.5\times10^2$ $5.4\times10^2$ $7.1\times10^1$ | | Wang et al. (2017) Wang et al. (2017) Wang et al. (2017) | Q Q Q | 80, 238 80, 239 80, 240 |
| MCM:ISOPDOH $C_5H_{10}O_2$ HBHXLSPUXXXICF-UHFFFAOYSA-N | $5.3\times10^2$ $7.4\times10^2$ $2.1\times10^2$ | | Wang et al. (2017) Wang et al. (2017) Wang et al. (2017) | Q Q Q | 80, 238 80, 239 80, 240 |
| MCM:C622OH $C_6H_{12}O_2$ VCGCHTQYUUAIGQ-UHFFFAOYSA-N | $1.0\times10^3$ $4.3\times10^3$ $7.6\times10^2$ | | Wang et al. (2017) Wang et al. (2017) Wang et al. (2017) | Q Q Q | 80, 238 80, 239 80, 240 |
| MCM:C624OH $C_6H_{12}O_2$ BZQZWOYZOVQJLT-UHFFFAOYSA-N | $1.0\times10^3$ $3.6\times10^3$ $8.7\times10^2$ | | Wang et al. (2017) Wang et al. (2017) Wang et al. (2017) | Q Q Q | 80, 238 80, 239 80, 240 |



Table A3.3: Polyols ($R(OH)_n$) (... continued)

| Substance<br>Formula<br>(Trivial Name)<br>[CAS Registry Number]<br>InChIKey | $H_s^{cp}$<br>(at $T^\ominus$)<br>$\left[\dfrac{\text{mol}}{\text{m}^3\,\text{Pa}}\right]$ | $\dfrac{\text{d}\ln H_s^{cp}}{\text{d}(1/T)}$<br><br>[K] | Reference | Type | Note |
|---|---|---|---|---|---|
| MCM:C720OH<br>$C_7H_{12}O_2$<br>JRLLENLHZCUEBJ-UHFFFAOYSA-N | $4.5\times10^3$<br>$8.7\times10^4$<br>$7.6\times10^4$ | | Wang et al. (2017)<br>Wang et al. (2017)<br>Wang et al. (2017) | Q<br>Q<br>Q | 80, 238<br>80, 239<br>80, 240 |
| MCM:LIMAOH<br>$C_{10}H_{18}O_2$<br>WKZWTZTZWGWEGE-UHFFFAOYSA-N | $5.1\times10^2$<br>$5.5\times10^3$<br>$1.9\times10^3$ | | Wang et al. (2017)<br>Wang et al. (2017)<br>Wang et al. (2017) | Q<br>Q<br>Q | 80, 238<br>80, 239<br>80, 240 |
| MCM:LIMCOH<br>$C_{10}H_{18}O_2$<br>ZJALAEQNHJQSTN-UHFFFAOYSA-N | $6.9\times10^2$<br>$4.3\times10^3$<br>$1.2\times10^3$ | | Wang et al. (2017)<br>Wang et al. (2017)<br>Wang et al. (2017) | Q<br>Q<br>Q | 80, 238<br>80, 239<br>80, 240 |
| (*E,E*)-8,10-dodecadien-1-ol<br>$C_{12}H_{22}O$<br>(codlemone)<br>[33956-49-9]<br>CSWBSLXBXRFNST-MQQKCMAXSA-N | 1.7 | | Ebert et al. (2023) | ? | 316 |
| MCM:BCAOH<br>$C_{15}H_{26}O_2$<br>XZTGVWRBZDBQLP-UHFFFAOYSA-N | $5.4\times10^2$<br>$1.2\times10^4$<br>$1.1\times10^4$ | | Wang et al. (2017)<br>Wang et al. (2017)<br>Wang et al. (2017) | Q<br>Q<br>Q | 80, 238<br>80, 239<br>80, 240 |
| MCM:BCCOH<br>$C_{15}H_{26}O_2$<br>ALNVAIGKARRJOT-UHFFFAOYSA-N | $6.8\times10^2$<br>$1.8\times10^4$<br>$5.9\times10^3$ | | Wang et al. (2017)<br>Wang et al. (2017)<br>Wang et al. (2017) | Q<br>Q<br>Q | 80, 238<br>80, 239<br>80, 240 |
| 1,2-dihydroxybenzene<br>$C_6H_4(OH)_2$<br>(pyrocatechol)<br>[120-80-9]<br>YCIMNLLNPGFGHC-UHFFFAOYSA-N | $8.2\times10^3$<br>$8.2\times10^3$<br>$1.8\times10^3$<br>$1.6\times10^2$<br>$4.5\times10^1$<br>$2.6\times10^3$<br>$1.4\times10^4$<br>$5.4\times10^3$<br>$4.4\times10^3$<br>$7.8\times10^3$<br>$1.6\times10^3$<br>$1.6\times10^5$<br>$1.2\times10^3$<br>$7.9\times10^2$<br><br> | <br><br><br><br><br><br><br><br><br><br><br><br><br><br>8300<br>7400 | Duchowicz et al. (2020)<br>HSDB (2015)<br>Mackay et al. (2006c)<br>Schüürmann (2000)<br>Mackay et al. (1995)<br>Duchowicz et al. (2020)<br>Wang et al. (2017)<br>Wang et al. (2017)<br>Wang et al. (2017)<br>Raventos-Duran et al. (2010)<br>Raventos-Duran et al. (2010)<br>Raventos-Duran et al. (2010)<br>Hilal et al. (2008)<br>Modarresi et al. (2007)<br>Kühne et al. (2005)<br>Kühne et al. (2005) | V<br>V<br>V<br>V<br>V<br>Q<br>Q<br>Q<br>Q<br>Q<br>Q<br>Q<br>Q<br>Q<br>Q<br>? | 186<br><br><br><br><br><br>80, 238<br>80, 239<br>80, 240<br>242, 243<br>244<br>245<br><br>67<br><br> |
| 1,3-dihydroxybenzene<br>$C_6H_4(OH)_2$<br>(resorcinol)<br>[108-46-3]<br>GHMLBKRAJCXXBS-UHFFFAOYSA-N | $1.0\times10^5$<br>$1.0\times10^5$<br>$8.5\times10^4$<br>$5.0\times10^3$<br>$6.4\times10^4$<br>$8.1\times10^4$<br>$1.4\times10^5$<br>$2.4\times10^4$ | <br><br><br><br><br>6300<br><br> | Duchowicz et al. (2020)<br>HSDB (2015)<br>Mackay et al. (2006c)<br>Schüürmann (2000)<br>Goldstein (1982)<br>Goldstein (1982)<br>Duchowicz et al. (2020)<br>Gharagheizi et al. (2012) | V<br>V<br>V<br>V<br>X<br>X<br>Q<br>Q | 186<br><br><br><br>446<br>298<br><br> |





Table A3.3: Polyols $(R(OH)_n)$ (... continued)

| Substance Formula (Trivial Name) [CAS Registry Number] InChIKey | $H_s^{cp}$ (at $T^{\ominus}$) $\left[\dfrac{\mathrm{mol}}{\mathrm{m^3\,Pa}}\right]$ | $\dfrac{\mathrm{d}\ln H_s^{cp}}{\mathrm{d}(1/T)}$ [K] | Reference | Type | Note |
|---|---|---|---|---|---|
| | $5.3\times10^4$ | | Hilal et al. (2008) | Q | |
| 1,4-dihydroxybenzene | $2.1\times10^5$ | | Duchowicz et al. (2020) | V | 186 |
| $C_6H_4(OH)_2$ | $2.6\times10^5$ | | HSDB (2015) | V | |
| (hydroquinone) | $2.5\times10^5$ | | Mackay et al. (2006c) | V | |
| [123-31-9] | $3.2\times10^4$ | | Schüürmann (2000) | V | |
| QIGBRXMKCJKVMJ-UHFFFAOYSA-N | $2.5\times10^5$ | | Mackay et al. (1995) | V | |
| | $2.6\times10^5$ | | Meylan and Howard (1991) | V | |
| | $7.6\times10^3$ | | Yaws (2003) | X | 258 |
| | $2.3\times10^5$ | | Dupeux et al. (2022) | Q | 259 |
| | $1.3\times10^5$ | | Duchowicz et al. (2020) | Q | |
| | $3.1\times10^4$ | | Raventos-Duran et al. (2010) | Q | 271, 243 |
| | $1.2\times10^5$ | | Raventos-Duran et al. (2010) | Q | 244 |
| | $1.6\times10^5$ | | Raventos-Duran et al. (2010) | Q | 245 |
| | $3.7\times10^4$ | | Hilal et al. (2008) | Q | |
| | | 8300 | Kühne et al. (2005) | Q | |
| | $1.7\times10^5$ | | Meylan and Howard (1991) | Q | |
| | | 7700 | Kühne et al. (2005) | ? | |
| 1,2,3-benzenetriol | $6.3\times10^4$ | | HSDB (2015) | V | |
| $C_6H_6O_3$ | $5.1\times10^6$ | | Gharagheizi et al. (2012) | Q | |
| (pyrogallic acid) [87-66-1] WQGWDDDVZFFDIG-UHFFFAOYSA-N | | | | | |
| 1,2-dihydroxy-3-methylbenzene | $8.1\times10^3$ | | Wang et al. (2017) | Q | 80, 238 |
| $C_7H_8O_2$ | $3.7\times10^3$ | | Wang et al. (2017) | Q | 80, 239 |
| [488-17-5] PGSWEKYNAOWQDF-UHFFFAOYSA-N | $7.4\times10^2$ | | Wang et al. (2017) | Q | 80, 240 |
| MCM:ECATECHOL | $7.4\times10^3$ | | Wang et al. (2017) | Q | 80, 238 |
| $C_8H_{10}O_2$ | $2.3\times10^3$ | | Wang et al. (2017) | Q | 80, 239 |
| UUCQGNWZASKXNN-UHFFFAOYSA-N | $3.8\times10^2$ | | Wang et al. (2017) | Q | 80, 240 |
| MCM:MXYCATECH | $4.8\times10^3$ | | Wang et al. (2017) | Q | 80, 238 |
| $C_8H_{10}O_2$ | $2.9\times10^3$ | | Wang et al. (2017) | Q | 80, 239 |
| YGLVLWAMIJMBPF-UHFFFAOYSA-N | $6.0\times10^2$ | | Wang et al. (2017) | Q | 80, 240 |
| MCM:OXYCATECH | $4.8\times10^3$ | | Wang et al. (2017) | Q | 80, 238 |
| $C_8H_{10}O_2$ | $3.6\times10^3$ | | Wang et al. (2017) | Q | 80, 239 |
| RYHGQTREHREIBC-UHFFFAOYSA-N | $3.1\times10^2$ | | Wang et al. (2017) | Q | 80, 240 |
| MCM:PXYCATECH | $4.8\times10^3$ | | Wang et al. (2017) | Q | 80, 238 |
| $C_8H_{10}O_2$ | $3.9\times10^3$ | | Wang et al. (2017) | Q | 80, 239 |
| RGUZWBOJHNWZOK-UHFFFAOYSA-N | $7.8\times10^1$ | | Wang et al. (2017) | Q | 80, 240 |
| MCM:IPCATECHOL | $6.8\times10^3$ | | Wang et al. (2017) | Q | 80, 238 |
| $C_9H_{12}O_2$ | $1.1\times10^3$ | | Wang et al. (2017) | Q | 80, 239 |
| XLZHGKDRKSKCAU-UHFFFAOYSA-N | $2.9\times10^2$ | | Wang et al. (2017) | Q | 80, 240 |



Table A3.3: Polyols ($R(OH)_n$) (... continued)

| Substance<br>Formula<br>(Trivial Name)<br>[CAS Registry Number]<br>InChIKey | $H_s^{cp}$<br>(at $T^{\ominus}$)<br>$\left[\dfrac{\mathrm{mol}}{\mathrm{m^3\,Pa}}\right]$ | $\dfrac{\mathrm{d}\ln H_s^{cp}}{\mathrm{d}(1/T)}$<br><br>[K] | Reference | Type | Note |
|---|---|---|---|---|---|
| MCM:METCATECH<br>$C_9H_{12}O_2$<br>IFERDVOUSBGAOD-UHFFFAOYSA-N | $4.4\times10^3$<br>$1.9\times10^3$<br>$3.3\times10^2$ | | Wang et al. (2017)<br>Wang et al. (2017)<br>Wang et al. (2017) | Q<br>Q<br>Q | 80, 238<br>80, 239<br>80, 240 |
| MCM:OETCATECH<br>$C_9H_{12}O_2$<br>QTXIMHKRKLVXRS-UHFFFAOYSA-N | $4.4\times10^3$<br>$2.2\times10^3$<br>$3.4\times10^2$ | | Wang et al. (2017)<br>Wang et al. (2017)<br>Wang et al. (2017) | Q<br>Q<br>Q | 80, 238<br>80, 239<br>80, 240 |
| MCM:PCATECHOL<br>$C_9H_{12}O_2$<br>GOZVFLWHGAXTPA-UHFFFAOYSA-N | $5.9\times10^3$<br>$1.5\times10^3$<br>$3.6\times10^2$ | | Wang et al. (2017)<br>Wang et al. (2017)<br>Wang et al. (2017) | Q<br>Q<br>Q | 80, 238<br>80, 239<br>80, 240 |
| MCM:PETCATECH<br>$C_9H_{12}O_2$<br>WVRWBYUJPWRFQO-UHFFFAOYSA-N | $4.4\times10^3$<br>$2.5\times10^3$<br>$4.3\times10^1$ | | Wang et al. (2017)<br>Wang et al. (2017)<br>Wang et al. (2017) | Q<br>Q<br>Q | 80, 238<br>80, 239<br>80, 240 |
| MCM:T123CATECH<br>$C_9H_{12}O_2$<br>DEIKGXRMQUHZJD-UHFFFAOYSA-N | $2.9\times10^3$<br>$4.6\times10^3$<br>$5.1\times10^2$ | | Wang et al. (2017)<br>Wang et al. (2017)<br>Wang et al. (2017) | Q<br>Q<br>Q | 80, 238<br>80, 239<br>80, 240 |
| MCM:T124CATECH<br>$C_9H_{12}O_2$<br>NZEZVJPYSAXNTR-UHFFFAOYSA-N | $2.9\times10^3$<br>$4.9\times10^3$<br>$4.8\times10^1$ | | Wang et al. (2017)<br>Wang et al. (2017)<br>Wang et al. (2017) | Q<br>Q<br>Q | 80, 238<br>80, 239<br>80, 240 |
| 2,3-dihydroxynaphthalene<br>$C_{10}H_8O_2$<br>[92-44-4]<br>JRNGUTKWMSBIBF-UHFFFAOYSA-N | $2.0\times10^5$ | | Ebert et al. (2023) | ? | 316 |
| hexylresorcinol<br>$C_{12}H_{18}O_2$<br>[136-77-6]<br>WFJIVOKAWHGMBH-UHFFFAOYSA-N | $3.8\times10^4$ | | HSDB (2015) | Q | 99 |
| 2,6-bis(1,1-dimethylethyl)phenol<br>$C_{14}H_{22}O$<br>[128-39-2]<br>DKCPKDPYUFEZCP-UHFFFAOYSA-N | 3.1 | | HSDB (2015) | Q | 99 |
| 4-(1-methyl-1-phenylethyl)phenol<br>$C_{15}H_{16}O$<br>[599-64-4]<br>QBDSZLJBMIMQRS-UHFFFAOYSA-N | $1.1\times10^2$ | | HSDB (2015) | Q | 447 |
| 2,2',3,3'-tetrahydro-3,3,3',3'-<br>tetramethyl-1,1'-spirobi(1H-<br>indene)-6,6'-diol<br><br>$C_{21}H_{24}O_2$<br>[1568-80-5]<br>SICLLPHPVFCNTJ-UHFFFAOYSA-N | $1.5\times10^6$<br><br><br><br>$1.0\times10^6$<br>$2.2\times10^6$<br>$8.2\times10^5$ | | Zhang et al. (2010)<br><br><br><br>Zhang et al. (2010)<br>Zhang et al. (2010)<br>Zhang et al. (2010) | Q<br><br><br><br>Q<br>Q<br>Q | 287, 288<br><br><br><br>287, 289<br>287, 290<br>287, 291 |



Table A3.3: Polyols $(R(OH)_n)$ (... continued)

| Substance Formula (Trivial Name) [CAS Registry Number] InChIKey | $H_s^{cp}$ (at $T^\ominus$) $\left[\dfrac{\text{mol}}{\text{m}^3\,\text{Pa}}\right]$ | $\dfrac{\text{d}\ln H_s^{cp}}{\text{d}(1/T)}$ [K] | Reference | Type | Note |
|---|---|---|---|---|---|
| 4,4'-(3,3,5-trimethylcyclohexane-1,1-diyl)diphenol | $4.4\times10^5$ | | Zhang et al. (2010) | Q | 287, 288 |
| $C_{21}H_{26}O_2$ | $3.2\times10^5$ | | Zhang et al. (2010) | Q | 287, 289 |
| [129188-99-4] | $2.7\times10^6$ | | Zhang et al. (2010) | Q | 287, 290 |
| UMPGNGRIGSEMTC-UHFFFAOYSA-N | $7.9\times10^5$ | | Zhang et al. (2010) | Q | 287, 291 |
| 3,3,3',3'-tetramethyl-1,1'-spirobi(indan)-5,5',6,6'-tetrol | $1.4\times10^{14}$ | | Zhang et al. (2010) | Q | 287, 288 |
| $C_{21}H_{24}O_4$ | $1.6\times10^{10}$ | | Zhang et al. (2010) | Q | 287, 289 |
| [77-08-7] | $2.0\times10^{11}$ | | Zhang et al. (2010) | Q | 287, 290 |
| POFMQEVZKZVAPQ-UHFFFAOYSA-N | $2.7\times10^{10}$ | | Zhang et al. (2010) | Q | 287, 291 |
| 9,9-bis(4-hydroxyphenyl)fluorene | $8.4\times10^8$ | | Zhang et al. (2010) | Q | 287, 288 |
| $C_{25}H_{18}O_2$ | $6.2\times10^7$ | | Zhang et al. (2010) | Q | 287, 289 |
| [3236-71-3] | $2.1\times10^8$ | | Zhang et al. (2010) | Q | 287, 290 |
| YWFPGFJLYRKYJZ-UHFFFAOYSA-N | $3.1\times10^9$ | | Zhang et al. (2010) | Q | 287, 291 |





### A3.4   Peroxides (ROOH) and peroxy radicals (ROO)

Table A3.4: Peroxides (ROOH) and peroxy radicals (ROO)

| Substance Formula (Trivial Name) [CAS Registry Number] InChIKey | $H_s^{cp}$ (at $T^{\ominus}$) $\left[\dfrac{\text{mol}}{\text{m}^3\,\text{Pa}}\right]$ | $\dfrac{\text{d}\ln H_s^{cp}}{\text{d}(1/T)}$ [K] | Reference | Type | Note |
|---|---|---|---|---|---|
| methyl hydroperoxide | 3.0 | 5300 | Burkholder et al. (2019) | L | |
| CH$_3$OOH | 3.0 | 5300 | Burkholder et al. (2015) | L | |
| (methyl peroxide) | 3.0 | 5200 | Brockbank (2013) | L | 1 |
| [3031-73-0] | 2.9 | 5200 | Warneck and Williams (2012) | L | |
| MEUKEBNAABNAEX-UHFFFAOYSA-N | 3.0 | 5300 | Sander et al. (2011) | L | |
| | 3.0 | 5300 | Sander et al. (2006) | L | |
| | 3.1 | 5300 | Staudinger and Roberts (2001) | L | |
| | 2.5 | 4400 | Li et al. (2004) | M | |
| | >6.9 | | Magi et al. (1997) | M | 448 |
| | $1.2\times10^1$ | | Sauer (1997) | M | 449 |
| | 3.1 | 5200 | O'Sullivan et al. (1996) | M | |
| | 3.0 | 5300 | Lind and Kok (1994) | M | 52 |
| | $1.0\times10^1$ | | Wang et al. (2017) | Q | 80, 238 |
| | $1.0\times10^1$ | | Wang et al. (2017) | Q | 80, 239 |
| | 6.2 | | Wang et al. (2017) | Q | 80, 240 |
| | 4.9 | | Raventos-Duran et al. (2010) | Q | 271, 243 |
| | 1.2 | | Raventos-Duran et al. (2010) | Q | 244 |
| | 1.6 | | Raventos-Duran et al. (2010) | Q | 245 |
| | $9.0\times10^{-1}$ | | Hilal et al. (2008) | Q | |
| | $1.3\times10^1$ | | Modarresi et al. (2007) | Q | 67 |
| | | 6200 | Kühne et al. (2005) | Q | |
| | | 5200 | Kühne et al. (2005) | ? | |
| ethyl hydroperoxide | 3.3 | 6000 | Burkholder et al. (2019) | L | |
| C$_2$H$_5$OOH | 3.3 | 6000 | Burkholder et al. (2015) | L | |
| (ethyl peroxide) | 3.3 | 6000 | Brockbank (2013) | L | 1 |
| [3031-74-1] | 3.3 | 6000 | Sander et al. (2011) | L | |
| ILHIHKRJJMKBEE-UHFFFAOYSA-N | $1.1\times10^1$ | | Sauer (1997) | M | 449 |
| | 3.3 | 6000 | O'Sullivan et al. (1996) | M | |
| | 5.0 | | Keshavarz et al. (2022) | Q | |
| | $1.0\times10^1$ | | Duchowicz et al. (2020) | Q | 184 |
| | 8.1 | | Wang et al. (2017) | Q | 80, 238 |
| | 8.0 | | Wang et al. (2017) | Q | 80, 239 |
| | 5.5 | | Wang et al. (2017) | Q | 80, 240 |
| | 3.9 | | Raventos-Duran et al. (2010) | Q | 271, 243 |
| | $9.9\times10^{-1}$ | | Raventos-Duran et al. (2010) | Q | 244 |
| | $9.9\times10^{-1}$ | | Raventos-Duran et al. (2010) | Q | 245 |
| | $5.8\times10^{-1}$ | | Hilal et al. (2008) | Q | |
| | | 6600 | Kühne et al. (2005) | Q | |
| | 3.4 | | Duchowicz et al. (2020) | ? | 185, 21 |
| | | 6000 | Kühne et al. (2005) | ? | |



Table A3.4: Peroxides (ROOH) and peroxy radicals (ROO) (. . . continued)

| Substance<br>Formula<br>(Trivial Name)<br>[CAS Registry Number]<br>InChIKey | $H_s^{cp}$<br>(at $T^{\ominus}$)<br>$\left[\dfrac{\text{mol}}{\text{m}^3\,\text{Pa}}\right]$ | $\dfrac{\text{d}\ln H_s^{cp}}{\text{d}(1/T)}$<br><br>[K] | Reference | Type | Note |
|---|---|---|---|---|---|
| hydroxymethyl hydroperoxide | $1.7\times10^4$ | 9900 | Burkholder et al. (2019) | L | |
| HOCH$_2$OOH | $1.7\times10^4$ | 9900 | Burkholder et al. (2015) | L | |
| (HMHP; HMP) | $1.7\times10^4$ | 9900 | Sander et al. (2011) | L | |
| [15932-89-5] | $1.7\times10^4$ | 9900 | Sander et al. (2006) | L | |
| NEZWFWIACBUQMN-UHFFFAOYSA-N | $1.6\times10^4$ | 10000 | Staudinger and Roberts (2001) | L | |
| | $1.6\times10^4$ | 9700 | O'Sullivan et al. (1996) | M | |
| | $1.6\times10^4$ | 10000 | Staffelbach and Kok (1993) | M | |
| | $4.7\times10^3$ | 1500 | Zhou and Lee (1992) | M | |
| | $1.6\times10^4$ | | Raventos-Duran et al. (2010) | Q | 242, 243 |
| | $1.6\times10^2$ | | Raventos-Duran et al. (2010) | Q | 244 |
| | $3.9\times10^4$ | | Raventos-Duran et al. (2010) | Q | 245 |
| | | 8600 | Kühne et al. (2005) | Q | |
| | | 10000 | Kühne et al. (2005) | ? | |
| bis-(hydroxymethyl)-peroxide | $>9.9\times10^4$ | | Staffelbach and Kok (1993) | M | |
| HOCH$_2$OOCH$_2$OH | $4.4\times10^3$ | 8400 | Zhou and Lee (1992) | M | |
| (BHMP) | | 9400 | Kühne et al. (2005) | Q | |
| [17088-73-2] | | 8500 | Kühne et al. (2005) | ? | |
| JLJXMZMKMRQOLN-UHFFFAOYSA-N | | | | | |
| *tert*-butyl hydroperoxide | 4.2 | | Wang et al. (2017) | Q | 80, 238 |
| C$_4$H$_{10}$O$_2$ | 1.4 | | Wang et al. (2017) | Q | 80, 239 |
| [75-91-2] | 3.6 | | Wang et al. (2017) | Q | 80, 240 |
| CIHOLLKRGTVIJN-UHFFFAOYSA-N | $6.2\times10^{-1}$ | | HSDB (2015) | Q | 99 |
| di-*tert*-butylperoxide | $8.2\times10^{-4}$ | | HSDB (2015) | Q | 99 |
| C$_8$H$_{18}$O$_2$ | $1.2\times10^{-4}$ | | Hilal et al. (2008) | Q | |
| [110-05-4] | $1.6\times10^{-2}$ | | Modarresi et al. (2007) | Q | 67 |
| LSXWFXONGKSEMY-UHFFFAOYSA-N | | | | | |
| 1-methyl-1-<br>phenylethylhydroperoxide | $2.1\times10^2$ | | Duchowicz et al. (2020) | V | 186 |
| C$_9$H$_{12}$O$_2$ | $2.1\times10^2$ | | HSDB (2015) | V | |
| [80-15-9] | $2.3\times10^1$ | | Duchowicz et al. (2020) | Q | |
| YQHLDYVWEZKEOX-UHFFFAOYSA-N | $1.3\times10^2$ | | Wang et al. (2017) | Q | 80, 238 |
| | $2.3\times10^1$ | | Wang et al. (2017) | Q | 80, 239 |
| | $2.3\times10^1$ | | Wang et al. (2017) | Q | 80, 240 |
| | 2.3 | | Hilal et al. (2008) | Q | |
| | $7.3\times10^1$ | | Modarresi et al. (2007) | Q | 67 |
| dicumyl peroxide | $2.2\times10^{-1}$ | | HSDB (2015) | Q | 99 |
| C$_{18}$H$_{22}$O$_2$ | | | | | |
| [80-43-3] | | | | | |
| XMNIXWIUMCBBBL-UHFFFAOYSA-N | | | | | |
| methylperoxy radical | $1.5\times10^{-1}$ | 3700 | Leriche et al. (2000) | E | 450 |
| CH$_3$OO | | | Lelieveld and Crutzen (1991) | E | 451 |
| [2143-58-0] | $5.9\times10^{-2}$ | 5600 | Jacob (1986) | E | 452 |
| WTFNSXYULBQCQV-UHFFFAOYSA-N | | | | | |





Table A3.4: Peroxides (ROOH) and peroxy radicals (ROO) (. . . continued)

| Substance Formula (Trivial Name) [CAS Registry Number] InChIKey | $H_s^{cp}$ (at $T^\ominus$) $\left[\dfrac{\mathrm{mol}}{\mathrm{m}^3\,\mathrm{Pa}}\right]$ | $\dfrac{\mathrm{d}\ln H_s^{cp}}{\mathrm{d}(1/T)}$ [K] | Reference | Type | Note |
|---|---|---|---|---|---|
| hydroxymethylperoxy radical HOCH$_2$OO [27828-51-9] OLHGCLQZCFQBGO-UHFFFAOYSA-N | $7.9\times10^2$ | 8200 | Leriche et al. (2000) | E | 450 |
| peroxyacetyl radical CH$_3$C(O)O$_2$ [36709-10-1] ZBQKPDHUDKSCRS-UHFFFAOYSA-N | $<9.9\times10^{-4}$ $<9.9\times10^{-4}$ $<9.9\times10^{-4}$ $<9.9\times10^{-4}$ $<9.9\times10^{-4}$ | | Burkholder et al. (2019) Burkholder et al. (2015) Sander et al. (2011) Sander et al. (2006) Villalta et al. (1996) | L L L L M | |
| MCM:ACO3H C$_3$H$_4$O$_3$ AZIQALWHRUQPHV-UHFFFAOYSA-N | $2.7\times10^2$ $1.4\times10^1$ $2.0\times10^{-1}$ | | Wang et al. (2017) Wang et al. (2017) Wang et al. (2017) | Q Q Q | 80, 238 80, 239 80, 240 |
| MCM:IC3H7OOH C$_3$H$_8$O$_2$ SGJUFIMCHSLMRJ-UHFFFAOYSA-N | 7.6 3.9 4.4 | | Wang et al. (2017) Wang et al. (2017) Wang et al. (2017) | Q Q Q | 80, 238 80, 239 80, 240 |
| MCM:NC3H7OOH C$_3$H$_8$O$_2$ TURGQPDWYFJEDY-UHFFFAOYSA-N | 6.8 4.7 3.0 | | Wang et al. (2017) Wang et al. (2017) Wang et al. (2017) | Q Q Q | 80, 238 80, 239 80, 240 |
| MCM:PERPROACID C$_3$H$_6$O$_3$ CZPZWMPYEINMCF-UHFFFAOYSA-N | $9.8\times10^1$ 5.3 $6.3\times10^{-2}$ | | Wang et al. (2017) Wang et al. (2017) Wang et al. (2017) | Q Q Q | 80, 238 80, 239 80, 240 |
| MCM:C3DBCO3H C$_4$H$_6$O$_3$ TXNBRZBEFMJQMW-UHFFFAOYSA-N | $3.2\times10^2$ $1.0\times10^1$ $5.4\times10^{-2}$ | | Wang et al. (2017) Wang et al. (2017) Wang et al. (2017) | Q Q Q | 80, 238 80, 239 80, 240 |
| MCM:IC4H9OOH C$_4$H$_{10}$O$_2$ FUHWWEDRJKHMKK-UHFFFAOYSA-N | 6.0 3.6 2.2 | | Wang et al. (2017) Wang et al. (2017) Wang et al. (2017) | Q Q Q | 80, 238 80, 239 80, 240 |
| MCM:MACO3H C$_4$H$_6$O$_3$ OELQSSWXRGADDE-UHFFFAOYSA-N | $1.8\times10^2$ 8.7 $1.6\times10^{-1}$ | | Wang et al. (2017) Wang et al. (2017) Wang et al. (2017) | Q Q Q | 80, 238 80, 239 80, 240 |
| IEPOX1CO3H C$_4$H$_6$O$_5$ DJPWYFXNRMHUNU-UHFFFAOYSA-N | $9.0\times10^2$ | 14000 | Wieser et al. (2023) | Q | 437 |
| MCM:NC4H9OOH C$_4$H$_{10}$O$_2$ AKUNSTOMHUXJOZ-UHFFFAOYSA-N | 5.3 3.4 3.2 | | Wang et al. (2017) Wang et al. (2017) Wang et al. (2017) | Q Q Q | 80, 238 80, 239 80, 240 |
| MCM:PERBUACID C$_4$H$_8$O$_3$ LBAYFEDWGHXMSM-UHFFFAOYSA-N | $7.8\times10^1$ 2.8 $4.1\times10^{-2}$ | | Wang et al. (2017) Wang et al. (2017) Wang et al. (2017) | Q Q Q | 80, 238 80, 239 80, 240 |





Table A3.4: Peroxides (ROOH) and peroxy radicals (ROO) (. . . continued)

| Substance Formula (Trivial Name) [CAS Registry Number] InChIKey | $H_s^{cp}$ (at $T^\ominus$) $\left[\dfrac{\text{mol}}{\text{m}^3\,\text{Pa}}\right]$ | $\dfrac{\text{d}\ln H_s^{cp}}{\text{d}(1/T)}$ [K] | Reference | Type | Note |
|---|---|---|---|---|---|
| MCM:PERIBUACID | $8.9\times10^1$ | | Wang et al. (2017) | Q | 80, 238 |
| $C_4H_8O_3$ | 3.2 | | Wang et al. (2017) | Q | 80, 239 |
| LVQKOPBJHBWELS-UHFFFAOYSA-N | $2.4\times10^{-2}$ | | Wang et al. (2017) | Q | 80, 240 |
| MCM:SC4H9OOH | 6.0 | | Wang et al. (2017) | Q | 80, 238 |
| $C_4H_{10}O_2$ | 2.4 | | Wang et al. (2017) | Q | 80, 239 |
| SPQMVUPFYWDFCB-UHFFFAOYSA-N | 2.4 | | Wang et al. (2017) | Q | 80, 240 |
| MCM:BUT2CO3H | $7.3\times10^1$ | | Wang et al. (2017) | Q | 80, 238 |
| $C_5H_{10}O_3$ | 2.0 | | Wang et al. (2017) | Q | 80, 239 |
| LWBFPIVRZGBOFS-UHFFFAOYSA-N | $1.8\times10^{-2}$ | | Wang et al. (2017) | Q | 80, 240 |
| MCM:C3ME3CO3H | $7.3\times10^1$ | | Wang et al. (2017) | Q | 80, 238 |
| $C_5H_{10}O_3$ | 2.3 | | Wang et al. (2017) | Q | 80, 239 |
| GTLLKMCVRPVGBK-UHFFFAOYSA-N | $2.8\times10^{-2}$ | | Wang et al. (2017) | Q | 80, 240 |
| MCM:IPEAOOH | 5.5 | | Wang et al. (2017) | Q | 80, 238 |
| $C_5H_{12}O_2$ | 2.6 | | Wang et al. (2017) | Q | 80, 239 |
| HIHRAMNMOMKJDG-UHFFFAOYSA-N | 1.9 | | Wang et al. (2017) | Q | 80, 240 |
| MCM:IPEBOOH | 5.6 | | Wang et al. (2017) | Q | 80, 238 |
| $C_5H_{12}O_2$ | 2.1 | | Wang et al. (2017) | Q | 80, 239 |
| VIUWCQFVAFABHL-UHFFFAOYSA-N | 2.1 | | Wang et al. (2017) | Q | 80, 240 |
| MCM:IPECOOH | 3.4 | | Wang et al. (2017) | Q | 80, 238 |
| $C_5H_{12}O_2$ | $9.8\times10^{-1}$ | | Wang et al. (2017) | Q | 80, 239 |
| XRXANEMIFVRKLN-UHFFFAOYSA-N | 1.9 | | Wang et al. (2017) | Q | 80, 240 |
| MCM:NEOPOOH | 3.4 | | Wang et al. (2017) | Q | 80, 238 |
| $C_5H_{12}O_2$ | 2.1 | | Wang et al. (2017) | Q | 80, 239 |
| UEQURRFROJBOLG-UHFFFAOYSA-N | 2.0 | | Wang et al. (2017) | Q | 80, 240 |
| MCM:PEAOOH | 4.7 | | Wang et al. (2017) | Q | 80, 238 |
| $C_5H_{12}O_2$ | 2.6 | | Wang et al. (2017) | Q | 80, 239 |
| KCHNMIKAMRQBHD-UHFFFAOYSA-N | 2.3 | | Wang et al. (2017) | Q | 80, 240 |
| MCM:PEBOOH | 5.5 | | Wang et al. (2017) | Q | 80, 238 |
| $C_5H_{12}O_2$ | 1.8 | | Wang et al. (2017) | Q | 80, 239 |
| XRIRVAYMWUMXBR-UHFFFAOYSA-N | 2.4 | | Wang et al. (2017) | Q | 80, 240 |
| MCM:PECOOH | 5.5 | | Wang et al. (2017) | Q | 80, 238 |
| $C_5H_{12}O_2$ | 1.7 | | Wang et al. (2017) | Q | 80, 239 |
| RLBQWRSRDVXRTJ-UHFFFAOYSA-N | 1.3 | | Wang et al. (2017) | Q | 80, 240 |
| MCM:PERPENACID | $6.0\times10^1$ | | Wang et al. (2017) | Q | 80, 238 |
| $C_5H_{10}O_3$ | 1.9 | | Wang et al. (2017) | Q | 80, 239 |
| UQGPCEVQKLOLLM-UHFFFAOYSA-N | $2.8\times10^{-2}$ | | Wang et al. (2017) | Q | 80, 240 |
| MCM:TBUTCO3H | $5.0\times10^1$ | | Wang et al. (2017) | Q | 80, 238 |
| $C_5H_{10}O_3$ | 1.8 | | Wang et al. (2017) | Q | 80, 239 |
| YVAACGXAZGGQSM-UHFFFAOYSA-N | $1.6\times10^{-2}$ | | Wang et al. (2017) | Q | 80, 240 |



Table A3.4: Peroxides (ROOH) and peroxy radicals (ROO) (. . . continued)

| Substance<br>Formula<br>(Trivial Name)<br>[CAS Registry Number]<br>InChIKey | $H_s^{cp}$<br>(at $T^{\ominus}$)<br>$\left[\dfrac{\text{mol}}{\text{m}^3\,\text{Pa}}\right]$ | $\dfrac{\text{d}\ln H_s^{cp}}{\text{d}(1/T)}$<br>[K] | Reference | Type | Note |
|---|---|---|---|---|---|
| C520OOH<br>$C_5H_{10}O_7$<br>GRPDYNZWDDUILX-UHFFFAOYSA-N | $5.3\times10^{11}$ | 22000 | Wieser et al. (2023) | Q | 437 |
| C518OOH<br>$C_5H_8O_4$<br>PQEISBQMLHJHJQ-UHFFFAOYSA-N | $4.1\times10^4$ | 14000 | Wieser et al. (2023) | Q | 437 |
| MCM:C54CO3H<br>$C_6H_{12}O_3$<br>JTSRVXQXWWOTER-UHFFFAOYSA-N | $5.6\times10^1$<br>1.4<br>$1.4\times10^{-2}$ | | Wang et al. (2017)<br>Wang et al. (2017)<br>Wang et al. (2017) | Q<br>Q<br>Q | 80, 238<br>80, 239<br>80, 240 |
| MCM:C5H11CO3H<br>$C_6H_{12}O_3$<br>NQUPKCJGWCPODR-UHFFFAOYSA-N | $5.6\times10^1$<br>1.4<br>$2.5\times10^{-2}$ | | Wang et al. (2017)<br>Wang et al. (2017)<br>Wang et al. (2017) | Q<br>Q<br>Q | 80, 238<br>80, 239<br>80, 240 |
| ROO6R7OOH<br>$C_6H_{12}O_4$<br>UPAWKBDHGPWUCG-UHFFFAOYSA-N | $3.3\times10^7$ | 14000 | Wieser et al. (2023) | Q | 437 |
| ROO6R6OOH<br>$C_6H_{12}O_5$<br>QZPUZDFJKAEVLL-UHFFFAOYSA-N | $2.6\times10^8$ | 14000 | Wieser et al. (2023) | Q | 437 |
| C624OOH<br>$C_6H_{12}O_6$<br>FUTSLQULSAKEJA-UHFFFAOYSA-N | $1.2\times10^{13}$ | 20000 | Wieser et al. (2023) | Q | 437 |
| MCM:CHEXOOH<br>$C_6H_{12}O_2$<br>FGGJBCRKSVGDPO-UHFFFAOYSA-N | $1.2\times10^1$<br>$1.4\times10^1$<br>$3.0\times10^1$ | | Wang et al. (2017)<br>Wang et al. (2017)<br>Wang et al. (2017) | Q<br>Q<br>Q | 80, 238<br>80, 239<br>80, 240 |
| MCM:HEXAOOH<br>$C_6H_{14}O_2$<br>RZICEOJUAFHYFO-UHFFFAOYSA-N | 3.8<br>2.1<br>2.2 | | Wang et al. (2017)<br>Wang et al. (2017)<br>Wang et al. (2017) | Q<br>Q<br>Q | 80, 238<br>80, 239<br>80, 240 |
| MCM:HEXBOOH<br>$C_6H_{14}O_2$<br>XWXUHAUZCICPHE-UHFFFAOYSA-N | $9.9\times10^{-1}$<br>4.4<br>1.4<br>1.8 | 7300 | Wieser et al. (2023)<br>Wang et al. (2017)<br>Wang et al. (2017)<br>Wang et al. (2017) | Q<br>Q<br>Q<br>Q | 437<br>80, 238<br>80, 239<br>80, 240 |
| MCM:HEXCOOH<br>$C_6H_{14}O_2$<br>NMNOLZYZNVYLRJ-UHFFFAOYSA-N | 4.4<br>1.4<br>1.1 | | Wang et al. (2017)<br>Wang et al. (2017)<br>Wang et al. (2017) | Q<br>Q<br>Q | 80, 238<br>80, 239<br>80, 240 |
| MCM:M22C3CO3H<br>$C_6H_{12}O_3$<br>RQLGMECUXRQENU-UHFFFAOYSA-N | $4.0\times10^1$<br>1.5<br>$2.1\times10^{-2}$ | | Wang et al. (2017)<br>Wang et al. (2017)<br>Wang et al. (2017) | Q<br>Q<br>Q | 80, 238<br>80, 239<br>80, 240 |
| MCM:M22C43OOH<br>$C_6H_{14}O_2$<br>XDWQBMWWYOVNSC-UHFFFAOYSA-N | 3.2<br>1.4<br>1.4 | | Wang et al. (2017)<br>Wang et al. (2017)<br>Wang et al. (2017) | Q<br>Q<br>Q | 80, 238<br>80, 239<br>80, 240 |





Table A3.4: Peroxides (ROOH) and peroxy radicals (ROO) (. . . continued)

| Substance Formula (Trivial Name) [CAS Registry Number] InChIKey | $H_s^{cp}$ (at $T^\ominus$) $\left[\dfrac{\text{mol}}{\text{m}^3\,\text{Pa}}\right]$ | $\dfrac{\text{d}\ln H_s^{cp}}{\text{d}(1/T)}$ [K] | Reference | Type | Note |
|---|---|---|---|---|---|
| MCM:M22C4OOH | 3.0 | | Wang et al. (2017) | Q | 80, 238 |
| C$_6$H$_{14}$O$_2$ | 2.0 | | Wang et al. (2017) | Q | 80, 239 |
| VWOCCXGIWJLCIT-UHFFFAOYSA-N | 3.6 | | Wang et al. (2017) | Q | 80, 240 |
| MCM:M23C43OOH | 3.2 | | Wang et al. (2017) | Q | 80, 238 |
| C$_6$H$_{14}$O$_2$ | 1.1 | | Wang et al. (2017) | Q | 80, 239 |
| UPTQBPSUWRBZRD-UHFFFAOYSA-N | 1.4 | | Wang et al. (2017) | Q | 80, 240 |
| MCM:M23C4OOH | 5.1 | | Wang et al. (2017) | Q | 80, 238 |
| C$_6$H$_{14}$O$_2$ | 2.6 | | Wang et al. (2017) | Q | 80, 239 |
| MCTQHWIBGOKRHO-UHFFFAOYSA-N | 1.6 | | Wang et al. (2017) | Q | 80, 240 |
| MCM:M2C43CO3H | $6.8\times10^1$ | | Wang et al. (2017) | Q | 80, 238 |
| C$_6$H$_{12}$O$_3$ | 1.8 | | Wang et al. (2017) | Q | 80, 239 |
| KGAQRBNLHLEGET-UHFFFAOYSA-N | $1.9\times10^{-2}$ | | Wang et al. (2017) | Q | 80, 240 |
| MCM:M2PEAOOH | 4.4 | | Wang et al. (2017) | Q | 80, 238 |
| C$_6$H$_{14}$O$_2$ | 2.1 | | Wang et al. (2017) | Q | 80, 239 |
| OTKKOLXSCMINFN-UHFFFAOYSA-N | 1.3 | | Wang et al. (2017) | Q | 80, 240 |
| MCM:M2PEBOOH | 5.1 | | Wang et al. (2017) | Q | 80, 238 |
| C$_6$H$_{14}$O$_2$ | 1.4 | | Wang et al. (2017) | Q | 80, 239 |
| XHHITPKXZQXJIC-UHFFFAOYSA-N | 2.2 | | Wang et al. (2017) | Q | 80, 240 |
| MCM:M2PECOOH | 5.1 | | Wang et al. (2017) | Q | 80, 238 |
| C$_6$H$_{14}$O$_2$ | 1.5 | | Wang et al. (2017) | Q | 80, 239 |
| JBJSMFBSVBRLAK-UHFFFAOYSA-N | 1.0 | | Wang et al. (2017) | Q | 80, 240 |
| MCM:M2PEDOOH | 3.0 | | Wang et al. (2017) | Q | 80, 238 |
| C$_6$H$_{14}$O$_2$ | $8.0\times10^{-1}$ | | Wang et al. (2017) | Q | 80, 239 |
| BZGMEGUFFDTCNP-UHFFFAOYSA-N | 1.3 | | Wang et al. (2017) | Q | 80, 240 |
| MCM:M33C3CO3H | $4.0\times10^1$ | | Wang et al. (2017) | Q | 80, 238 |
| C$_6$H$_{12}$O$_3$ | 1.2 | | Wang et al. (2017) | Q | 80, 239 |
| IHLWWYDUPGXTHJ-UHFFFAOYSA-N | $8.9\times10^{-2}$ | | Wang et al. (2017) | Q | 80, 240 |
| MCM:M33C4OOH | 3.0 | | Wang et al. (2017) | Q | 80, 238 |
| C$_6$H$_{14}$O$_2$ | 1.7 | | Wang et al. (2017) | Q | 80, 239 |
| OUMLJDCULPJYHU-UHFFFAOYSA-N | 1.8 | | Wang et al. (2017) | Q | 80, 240 |
| MCM:M3C4CO3H | $5.6\times10^1$ | | Wang et al. (2017) | Q | 80, 238 |
| C$_6$H$_{12}$O$_3$ | 1.7 | | Wang et al. (2017) | Q | 80, 239 |
| CHSVHHKRTQKZCN-UHFFFAOYSA-N | $2.3\times10^{-2}$ | | Wang et al. (2017) | Q | 80, 240 |
| MCM:M3PEAOOH | 4.4 | | Wang et al. (2017) | Q | 80, 238 |
| C$_6$H$_{14}$O$_2$ | 2.5 | | Wang et al. (2017) | Q | 80, 239 |
| AOZFMIRWQXSLHO-UHFFFAOYSA-N | 1.9 | | Wang et al. (2017) | Q | 80, 240 |
| MCM:M3PEBOOH | 5.1 | | Wang et al. (2017) | Q | 80, 238 |
| C$_6$H$_{14}$O$_2$ | 1.9 | | Wang et al. (2017) | Q | 80, 239 |
| XCSSHHOKPSJZJS-UHFFFAOYSA-N | 1.9 | | Wang et al. (2017) | Q | 80, 240 |



Table A3.4: Peroxides (ROOH) and peroxy radicals (ROO) (. . . continued)

| Substance Formula (Trivial Name) [CAS Registry Number] InChIKey | $H_s^{cp}$ (at $T^{\ominus}$) $\left[\dfrac{\mathrm{mol}}{\mathrm{m^3\,Pa}}\right]$ | $\dfrac{\mathrm{d}\ln H_s^{cp}}{\mathrm{d}(1/T)}$ [K] | Reference | Type | Note |
|---|---|---|---|---|---|
| MCM:M3PECOOH $C_6H_{14}O_2$ BWMWVYKKWYNWMF-UHFFFAOYSA-N | 3.0 $7.8\times10^{-1}$ 1.1 | | Wang et al. (2017) Wang et al. (2017) Wang et al. (2017) | Q Q Q | 80, 238 80, 239 80, 240 |
| ROO6R5OOH $C_7H_{12}O_6$ RAASFOSVQCCDRJ-UHFFFAOYSA-N | $4.6\times10^7$ | 16000 | Wieser et al. (2023) | Q | 437 |
| MCM:C6H13CO3H $C_7H_{14}O_3$ GLAYRDYYNSCSPO-UHFFFAOYSA-N | $4.4\times10^1$ 1.1 $2.2\times10^{-2}$ | | Wang et al. (2017) Wang et al. (2017) Wang et al. (2017) | Q Q Q | 80, 238 80, 239 80, 240 |
| MCM:HEPTOOH $C_7H_{16}O_2$ RCPMMXGHMBPBLI-UHFFFAOYSA-N | 3.6 1.1 $9.3\times10^{-1}$ | | Wang et al. (2017) Wang et al. (2017) Wang et al. (2017) | Q Q Q | 80, 238 80, 239 80, 240 |
| MCM:M2HEXAOOH $C_7H_{16}O_2$ MZRXBNCEXTUOSL-UHFFFAOYSA-N | 4.1 1.2 2.2 | | Wang et al. (2017) Wang et al. (2017) Wang et al. (2017) | Q Q Q | 80, 238 80, 239 80, 240 |
| MCM:M2HEXBOOH $C_7H_{16}O_2$ VERXWMXNIBWXFV-UHFFFAOYSA-N | 2.5 $6.6\times10^{-1}$ 1.1 | | Wang et al. (2017) Wang et al. (2017) Wang et al. (2017) | Q Q Q | 80, 238 80, 239 80, 240 |
| MCM:M3HEXAOOH $C_7H_{16}O_2$ SLXXGGKZGKGOKQ-UHFFFAOYSA-N | 4.1 1.4 1.2 | | Wang et al. (2017) Wang et al. (2017) Wang et al. (2017) | Q Q Q | 80, 238 80, 239 80, 240 |
| MCM:M3HEXBOOH $C_7H_{16}O_2$ YDBMAIVJCSJSKQ-UHFFFAOYSA-N | 2.5 $6.8\times10^{-1}$ 1.1 | | Wang et al. (2017) Wang et al. (2017) Wang et al. (2017) | Q Q Q | 80, 238 80, 239 80, 240 |
| 1-methylhexyl hydroperoxide $C_7H_{16}O_2$ (C7H15O2H) [762-46-9] FWELUXZVATZEMI-UHFFFAOYSA-N | $7.7\times10^{-1}$ | 7600 | Wieser et al. (2023) | Q | 437 |
| C7OHOOH $C_7H_{16}O_3$ KHWIBANUCNWWGW-UHFFFAOYSA-N | $4.1\times10^3$ | 12000 | Wieser et al. (2023) | Q | 437 |
| MCM:C8BCOOH $C_8H_{14}O_2$ TWXDGALDAWYAQL-UHFFFAOYSA-N | $2.7\times10^1$ 7.4 $1.1\times10^1$ | | Wang et al. (2017) Wang et al. (2017) Wang et al. (2017) | Q Q Q | 80, 238 80, 239 80, 240 |
| MCM:OCTOOH $C_8H_{18}O_2$ PXUMFRDGKQLDGL-UHFFFAOYSA-N | 2.8 $9.8\times10^{-1}$ 1.1 | | Wang et al. (2017) Wang et al. (2017) Wang et al. (2017) | Q Q Q | 80, 238 80, 239 80, 240 |



Table A3.4: Peroxides (ROOH) and peroxy radicals (ROO) (. . . continued)

| Substance Formula (Trivial Name) [CAS Registry Number] InChIKey | $H_s^{cp}$ (at $T^\ominus$) $\left[\dfrac{\text{mol}}{\text{m}^3\,\text{Pa}}\right]$ | $\dfrac{\mathrm{d}\ln H_s^{cp}}{\mathrm{d}(1/T)}$ [K] | Reference | Type | Note |
|---|---|---|---|---|---|
| 1-methylheptl hydroperoxide $C_8H_{18}O_2$ (C8H17O2H) NAXZMRYIZGEALQ-UHFFFAOYSA-N | $5.8\times10^{-1}$ | 8000 | Wieser et al. (2023) | Q | 437 |
| C8OHOOH $C_8H_{18}O_3$ QQTSMKQYSKGCOU-UHFFFAOYSA-N | $3.2\times10^3$ | 12000 | Wieser et al. (2023) | Q | 437 |
| MCM:NONOOH $C_9H_{20}O_2$ BXPIMWLPPYUCSQ-UHFFFAOYSA-N | 2.6 $8.3\times10^{-1}$ $9.6\times10^{-1}$ | | Wang et al. (2017) Wang et al. (2017) Wang et al. (2017) | Q Q Q | 80, 238 80, 239 80, 240 |
| MCM:DECOOH $C_{10}H_{22}O_2$ MPMVWDJTKHXCAN-UHFFFAOYSA-N | 2.0 $7.3\times10^{-1}$ 1.2 | | Wang et al. (2017) Wang et al. (2017) Wang et al. (2017) | Q Q Q | 80, 238 80, 239 80, 240 |
| C5H112O2H $C_{10}H_{22}O_4$ AXIHYONNAPPFSO-UHFFFAOYSA-N | 1.4 | 7000 | Wieser et al. (2023) | Q | 437 |
| LIMAB15OOH2 $C_{10}H_{20}O_6$ IRUZCTPAAZATQM-UHFFFAOYSA-N | $2.3\times10^{12}$ | 20000 | Wieser et al. (2023) | Q | 437 |
| RO5R1O2H $C_{10}H_{18}O_4$ NSOHXIGSWKLSHV-UHFFFAOYSA-N | $9.9\times10^5$ | 17000 | Wieser et al. (2023) | Q | 437 |
| RO5R2O2H $C_{10}H_{18}O_5$ MFXJOKCSTDMZRM-UHFFFAOYSA-N | $1.8\times10^7$ | 19000 | Wieser et al. (2023) | Q | 437 |
| ROO6R1OOH $C_{10}H_{18}O_5$ GBBPAXKBVMQHIG-UHFFFAOYSA-N | $6.8\times10^4$ | 16000 | Wieser et al. (2023) | Q | 437 |
| RO5R3O2H $C_{10}H_{18}O_6$ IEPFJXDIWPQIRC-UHFFFAOYSA-N | $7.7\times10^7$ | 22000 | Wieser et al. (2023) | Q | 437 |
| MCM:UDECOOH $C_{11}H_{24}O_2$ PPEGYVQDERYKNI-UHFFFAOYSA-N | 1.6 $6.5\times10^{-1}$ $9.6\times10^{-1}$ | | Wang et al. (2017) Wang et al. (2017) Wang et al. (2017) | Q Q Q | 80, 238 80, 239 80, 240 |
| MCM:DDECOOH $C_{12}H_{26}O_2$ ONWAGJMEBPWHEG-UHFFFAOYSA-N | 1.5 $5.8\times10^{-1}$ $8.3\times10^{-1}$ | | Wang et al. (2017) Wang et al. (2017) Wang et al. (2017) | Q Q Q | 80, 238 80, 239 80, 240 |
| MCM:C6H5OOH $C_6H_6O_2$ JYINMLPNDRBKKZ-UHFFFAOYSA-N | $3.2\times10^2$ $8.9\times10^1$ $3.0\times10^1$ | | Wang et al. (2017) Wang et al. (2017) Wang et al. (2017) | Q Q Q | 80, 238 80, 239 80, 240 |



Table A3.4: Peroxides (ROOH) and peroxy radicals (ROO) (...continued)

| Substance Formula (Trivial Name) [CAS Registry Number] InChIKey | $H_s^{cp}$ (at $T^\ominus$) $\left[\dfrac{\mathrm{mol}}{\mathrm{m}^3\,\mathrm{Pa}}\right]$ | $\dfrac{\mathrm{d}\ln H_s^{cp}}{\mathrm{d}(1/T)}$ [K] | Reference | Type | Note |
|---|---|---|---|---|---|
| MCM:C6H5CH2OOH | $2.6\times10^2$ | | Wang et al. (2017) | Q | 80, 238 |
| $C_7H_8O_2$ | $9.1\times10^1$ | | Wang et al. (2017) | Q | 80, 239 |
| YVJRCWCFDJYONJ-UHFFFAOYSA-N | $1.4\times10^2$ | | Wang et al. (2017) | Q | 80, 240 |
| MCM:C6H5CO3H | $3.7\times10^3$ | | Wang et al. (2017) | Q | 80, 238 |
| $C_7H_6O_3$ | $6.2\times10^1$ | | Wang et al. (2017) | Q | 80, 239 |
| XCRBXWCUXJNEFX-UHFFFAOYSA-N | $2.4$ | | Wang et al. (2017) | Q | 80, 240 |
| MCM:MXYL1OOH | $1.9\times10^2$ | | Wang et al. (2017) | Q | 80, 238 |
| $C_7H_8O_2$ | $7.8\times10^1$ | | Wang et al. (2017) | Q | 80, 239 |
| VCTGBWYTEXZDNM-UHFFFAOYSA-N | $3.0\times10^1$ | | Wang et al. (2017) | Q | 80, 240 |
| MCM:OXYL1OOH | $1.9\times10^2$ | | Wang et al. (2017) | Q | 80, 238 |
| $C_7H_8O_2$ | $1.0\times10^2$ | | Wang et al. (2017) | Q | 80, 239 |
| OCSKWYCRTWOMLG-UHFFFAOYSA-N | $2.5\times10^1$ | | Wang et al. (2017) | Q | 80, 240 |
| MCM:PXYL1OOH | $1.9\times10^2$ | | Wang et al. (2017) | Q | 80, 238 |
| $C_7H_8O_2$ | $9.1\times10^1$ | | Wang et al. (2017) | Q | 80, 239 |
| ZCUUEQJCOSBJOR-UHFFFAOYSA-N | $2.6\times10^1$ | | Wang et al. (2017) | Q | 80, 240 |
| MCM:C6H5C2CO3H | $3.1\times10^3$ | | Wang et al. (2017) | Q | 80, 238 |
| $C_8H_8O_3$ | $6.3\times10^1$ | | Wang et al. (2017) | Q | 80, 239 |
| BXGXGTXWGGOFSP-UHFFFAOYSA-N | $2.7$ | | Wang et al. (2017) | Q | 80, 240 |
| MCM:C6H5C2OOH | $2.3\times10^2$ | | Wang et al. (2017) | Q | 80, 238 |
| $C_8H_{10}O_2$ | $6.9\times10^1$ | | Wang et al. (2017) | Q | 80, 239 |
| VZQOBPXGQJXYGY-UHFFFAOYSA-N | $1.7\times10^2$ | | Wang et al. (2017) | Q | 80, 240 |
| MCM:DM123OOH | $1.1\times10^2$ | | Wang et al. (2017) | Q | 80, 238 |
| $C_8H_{10}O_2$ | $1.3\times10^2$ | | Wang et al. (2017) | Q | 80, 239 |
| FPXURWFXMHYDGM-UHFFFAOYSA-N | $2.6\times10^1$ | | Wang et al. (2017) | Q | 80, 240 |
| MCM:DM124OOH | $1.1\times10^2$ | | Wang et al. (2017) | Q | 80, 238 |
| $C_8H_{10}O_2$ | $1.0\times10^2$ | | Wang et al. (2017) | Q | 80, 239 |
| UQJVEYMRRMOLPI-UHFFFAOYSA-N | $3.2\times10^1$ | | Wang et al. (2017) | Q | 80, 240 |
| MCM:DMPHOOH | $1.1\times10^2$ | | Wang et al. (2017) | Q | 80, 238 |
| $C_8H_{10}O_2$ | $6.5\times10^1$ | | Wang et al. (2017) | Q | 80, 239 |
| AGQLZSPEKWKXRP-UHFFFAOYSA-N | $2.8\times10^1$ | | Wang et al. (2017) | Q | 80, 240 |
| MCM:EBENZOOH | $1.7\times10^2$ | | Wang et al. (2017) | Q | 80, 238 |
| $C_8H_{10}O_2$ | $6.3\times10^1$ | | Wang et al. (2017) | Q | 80, 239 |
| SULLUHFYVRLICT-UHFFFAOYSA-N | $2.0\times10^1$ | | Wang et al. (2017) | Q | 80, 240 |
| MCM:MXY1OOH | $1.1\times10^2$ | | Wang et al. (2017) | Q | 80, 238 |
| $C_8H_{10}O_2$ | $1.1\times10^2$ | | Wang et al. (2017) | Q | 80, 239 |
| LDNGPDBEDBLXJS-UHFFFAOYSA-N | $2.3\times10^1$ | | Wang et al. (2017) | Q | 80, 240 |
| MCM:MXYLCO3H | $2.2\times10^3$ | | Wang et al. (2017) | Q | 80, 238 |
| $C_8H_8O_3$ | $5.0\times10^1$ | | Wang et al. (2017) | Q | 80, 239 |
| INFFZTZRGNVBOC-UHFFFAOYSA-N | $2.7\times10^{-1}$ | | Wang et al. (2017) | Q | 80, 240 |





Table A3.4: Peroxides (ROOH) and peroxy radicals (ROO) (...continued)

| Substance Formula (Trivial Name) [CAS Registry Number] InChIKey | $H_s^{cp}$ (at $T^{\ominus}$) $\left[\dfrac{\mathrm{mol}}{\mathrm{m^3\,Pa}}\right]$ | $\dfrac{\mathrm{d}\ln H_s^{cp}}{\mathrm{d}(1/T)}$ [K] | Reference | Type | Note |
|---|---|---|---|---|---|
| MCM:MXYLOOH | $1.7\times10^2$ | | Wang et al. (2017) | Q | 80, 238 |
| $C_8H_{10}O_2$ | $8.3\times10^1$ | | Wang et al. (2017) | Q | 80, 239 |
| UTKFWOZXPFIBPQ-UHFFFAOYSA-N | $7.3\times10^1$ | | Wang et al. (2017) | Q | 80, 240 |
| MCM:OXYLCO3H | $2.2\times10^3$ | | Wang et al. (2017) | Q | 80, 238 |
| $C_8H_8O_3$ | $6.2\times10^1$ | | Wang et al. (2017) | Q | 80, 239 |
| DNEXRQSSNZPAJJ-UHFFFAOYSA-N | $2.0\times10^{-1}$ | | Wang et al. (2017) | Q | 80, 240 |
| MCM:OXYLOOH | $1.7\times10^2$ | | Wang et al. (2017) | Q | 80, 238 |
| $C_8H_{10}O_2$ | $9.3\times10^1$ | | Wang et al. (2017) | Q | 80, 239 |
| JTJQZHCBMBJJMV-UHFFFAOYSA-N | $6.0\times10^1$ | | Wang et al. (2017) | Q | 80, 240 |
| MCM:PXY1OOH | $1.1\times10^2$ | | Wang et al. (2017) | Q | 80, 238 |
| $C_8H_{10}O_2$ | $1.1\times10^2$ | | Wang et al. (2017) | Q | 80, 239 |
| QBGYVNGBYXOUAG-UHFFFAOYSA-N | $2.1\times10^1$ | | Wang et al. (2017) | Q | 80, 240 |
| MCM:PXYLCO3H | $2.2\times10^3$ | | Wang et al. (2017) | Q | 80, 238 |
| $C_8H_8O_3$ | $5.6\times10^1$ | | Wang et al. (2017) | Q | 80, 239 |
| IFDPVSBDNAQBRQ-UHFFFAOYSA-N | $7.1\times10^{-1}$ | | Wang et al. (2017) | Q | 80, 240 |
| MCM:PXYLOOH | $1.7\times10^2$ | | Wang et al. (2017) | Q | 80, 238 |
| $C_8H_{10}O_2$ | $9.8\times10^1$ | | Wang et al. (2017) | Q | 80, 239 |
| VCEHMDXCOGIOJJ-UHFFFAOYSA-N | $8.1\times10^1$ | | Wang et al. (2017) | Q | 80, 240 |
| MCM:EMPHOOH | $1.0\times10^2$ | | Wang et al. (2017) | Q | 80, 238 |
| $C_9H_{12}O_2$ | $4.0\times10^1$ | | Wang et al. (2017) | Q | 80, 239 |
| MKWUSWBHCPKVBI-UHFFFAOYSA-N | $2.5\times10^1$ | | Wang et al. (2017) | Q | 80, 240 |
| MCM:ETOLOOH | $1.6\times10^2$ | | Wang et al. (2017) | Q | 80, 238 |
| $C_9H_{12}O_2$ | $5.0\times10^1$ | | Wang et al. (2017) | Q | 80, 239 |
| DWPOLKIWKCGPAM-UHFFFAOYSA-N | $4.3\times10^1$ | | Wang et al. (2017) | Q | 80, 240 |
| MCM:IPBENZOOH | $1.6\times10^2$ | | Wang et al. (2017) | Q | 80, 238 |
| $C_9H_{12}O_2$ | $5.0\times10^1$ | | Wang et al. (2017) | Q | 80, 239 |
| CZNJIWCDRJWKLG-UHFFFAOYSA-N | $1.7\times10^1$ | | Wang et al. (2017) | Q | 80, 240 |
| MCM:MET1OOH | $1.0\times10^2$ | | Wang et al. (2017) | Q | 80, 238 |
| $C_9H_{12}O_2$ | $6.5\times10^1$ | | Wang et al. (2017) | Q | 80, 239 |
| HUAUUZDXRNWOSS-UHFFFAOYSA-N | $1.9\times10^1$ | | Wang et al. (2017) | Q | 80, 240 |
| MCM:OET1OOH | $1.0\times10^2$ | | Wang et al. (2017) | Q | 80, 238 |
| $C_9H_{12}O_2$ | $7.4\times10^1$ | | Wang et al. (2017) | Q | 80, 239 |
| KVZSMDLRNLVTLK-UHFFFAOYSA-N | $1.9\times10^1$ | | Wang et al. (2017) | Q | 80, 240 |
| MCM:PBENZOOH | $1.4\times10^2$ | | Wang et al. (2017) | Q | 80, 238 |
| $C_9H_{12}O_2$ | $4.9\times10^1$ | | Wang et al. (2017) | Q | 80, 239 |
| BFUPNGCSYHWOOH-UHFFFAOYSA-N | $1.8\times10^1$ | | Wang et al. (2017) | Q | 80, 240 |
| MCM:PET1OOH | $1.0\times10^2$ | | Wang et al. (2017) | Q | 80, 238 |
| $C_9H_{12}O_2$ | $6.6\times10^1$ | | Wang et al. (2017) | Q | 80, 239 |
| IKLGQZOVXYBNEV-UHFFFAOYSA-N | $1.7\times10^1$ | | Wang et al. (2017) | Q | 80, 240 |





Table A3.4: Peroxides (ROOH) and peroxy radicals (ROO) (...continued)

| Substance Formula (Trivial Name) [CAS Registry Number] InChIKey | $H_s^{cp}$ (at $T^{\ominus}$) $\left[\dfrac{\mathrm{mol}}{\mathrm{m}^3\,\mathrm{Pa}}\right]$ | $\dfrac{\mathrm{d}\ln H_s^{cp}}{\mathrm{d}(1/T)}$ [K] | Reference | Type | Note |
|---|---|---|---|---|---|
| MCM:PHC3OOH | $2.1\times10^2$ | | Wang et al. (2017) | Q | 80, 238 |
| $C_9H_{12}O_2$ | $3.6\times10^1$ | | Wang et al. (2017) | Q | 80, 239 |
| JYLUDNGUBXOJPX-UHFFFAOYSA-N | $5.4\times10^1$ | | Wang et al. (2017) | Q | 80, 240 |
| MCM:TM123BCO3H | $1.4\times10^3$ | | Wang et al. (2017) | Q | 80, 238 |
| $C_9H_{10}O_3$ | $7.3\times10^1$ | | Wang et al. (2017) | Q | 80, 239 |
| NQSDUJYDMWXHMR-UHFFFAOYSA-N | $3.4\times10^{-1}$ | | Wang et al. (2017) | Q | 80, 240 |
| MCM:TM123BOOH | $1.0\times10^2$ | | Wang et al. (2017) | Q | 80, 238 |
| $C_9H_{12}O_2$ | $1.2\times10^2$ | | Wang et al. (2017) | Q | 80, 239 |
| HSKYDUVZGAHSEO-UHFFFAOYSA-N | $1.6\times10^2$ | | Wang et al. (2017) | Q | 80, 240 |
| MCM:TM123OOH | $6.5\times10^1$ | | Wang et al. (2017) | Q | 80, 238 |
| $C_9H_{12}O_2$ | $1.6\times10^2$ | | Wang et al. (2017) | Q | 80, 239 |
| QIIZMGNSXSJCNA-UHFFFAOYSA-N | $5.4\times10^1$ | | Wang et al. (2017) | Q | 80, 240 |
| MCM:TM124BCO3H | $1.4\times10^3$ | | Wang et al. (2017) | Q | 80, 238 |
| $C_9H_{10}O_3$ | $6.0\times10^1$ | | Wang et al. (2017) | Q | 80, 239 |
| HPLBJCPAWPQKDU-UHFFFAOYSA-N | $4.8\times10^{-1}$ | | Wang et al. (2017) | Q | 80, 240 |
| MCM:TM124BOOH | $1.0\times10^2$ | | Wang et al. (2017) | Q | 80, 238 |
| $C_9H_{12}O_2$ | $1.1\times10^2$ | | Wang et al. (2017) | Q | 80, 239 |
| ZUEBLWOBXMLPMV-UHFFFAOYSA-N | $7.4\times10^1$ | | Wang et al. (2017) | Q | 80, 240 |
| MCM:TM124OOH | $6.5\times10^1$ | | Wang et al. (2017) | Q | 80, 238 |
| $C_9H_{12}O_2$ | $1.3\times10^2$ | | Wang et al. (2017) | Q | 80, 239 |
| DSGKIBNJTLHQCN-UHFFFAOYSA-N | $2.3\times10^1$ | | Wang et al. (2017) | Q | 80, 240 |
| MCM:TMBCO3H | $1.4\times10^3$ | | Wang et al. (2017) | Q | 80, 238 |
| $C_9H_{10}O_3$ | $3.9\times10^1$ | | Wang et al. (2017) | Q | 80, 239 |
| APDYULOMODUXLL-UHFFFAOYSA-N | $2.7\times10^{-1}$ | | Wang et al. (2017) | Q | 80, 240 |
| MCM:TMBOOH | $1.0\times10^2$ | | Wang et al. (2017) | Q | 80, 238 |
| $C_9H_{12}O_2$ | $6.9\times10^1$ | | Wang et al. (2017) | Q | 80, 239 |
| GOKTXPLVCSLYGI-UHFFFAOYSA-N | $1.2\times10^2$ | | Wang et al. (2017) | Q | 80, 240 |
| MCM:DM35EBOOH | $9.1\times10^1$ | | Wang et al. (2017) | Q | 80, 238 |
| $C_{10}H_{14}O_2$ | $3.6\times10^1$ | | Wang et al. (2017) | Q | 80, 239 |
| NSJZLSFVHPJVDL-UHFFFAOYSA-N | $4.9\times10^1$ | | Wang et al. (2017) | Q | 80, 240 |
| MCM:EMPHCO3H | $1.0\times10^3$ | | Wang et al. (2017) | Q | 80, 238 |
| $C_{10}H_{12}O_3$ | $2.3\times10^1$ | | Wang et al. (2017) | Q | 80, 239 |
| GYXFVTDASBFNQD-UHFFFAOYSA-N | $2.4\times10^{-1}$ | | Wang et al. (2017) | Q | 80, 240 |
| MCM:DE35TOOH | $7.6\times10^1$ | | Wang et al. (2017) | Q | 80, 238 |
| $C_{11}H_{16}O_2$ | $2.4\times10^1$ | | Wang et al. (2017) | Q | 80, 239 |
| HPTJZRKCFIXEOP-UHFFFAOYSA-N | $3.4\times10^1$ | | Wang et al. (2017) | Q | 80, 240 |
| MCM:HOCH2CO3H | $1.6\times10^4$ | | Wang et al. (2017) | Q | 80, 238 |
| $C_2H_4O_4$ | $1.9\times10^2$ | | Wang et al. (2017) | Q | 80, 239 |
| IUEZWLCUORJBDZ-UHFFFAOYSA-N | $8.5\times10^2$ | | Wang et al. (2017) | Q | 80, 240 |



Table A3.4: Peroxides (ROOH) and peroxy radicals (ROO) (...continued)

| Substance Formula (Trivial Name) [CAS Registry Number] InChIKey | $H_s^{cp}$ (at $T^{\ominus}$) $\left[\dfrac{\mathrm{mol}}{\mathrm{m^3\,Pa}}\right]$ | $\dfrac{\mathrm{d}\ln H_s^{cp}}{\mathrm{d}(1/T)}$ [K] | Reference | Type | Note |
|---|---|---|---|---|---|
| MCM:HYETHO2H | $2.8\times10^4$ | | Wang et al. (2017) | Q | 80, 238 |
| $C_2H_6O_3$ | $2.9\times10^3$ | | Wang et al. (2017) | Q | 80, 239 |
| FKPAKAOEHCFKLE-UHFFFAOYSA-N | $1.0\times10^4$ | | Wang et al. (2017) | Q | 80, 240 |
| MCM:C2OHOCOOH | $1.8\times10^7$ | | Wang et al. (2017) | Q | 80, 238 |
| $C_3H_6O_5$ | $2.2\times10^5$ | | Wang et al. (2017) | Q | 80, 239 |
| GTTVGDRVLZEEEW-UHFFFAOYSA-N | $3.3\times10^3$ | | Wang et al. (2017) | Q | 80, 240 |
| MCM:C3DIOLOOH | $2.8\times10^7$ | | Wang et al. (2017) | Q | 80, 238 |
| $C_3H_8O_4$ | $1.1\times10^6$ | | Wang et al. (2017) | Q | 80, 239 |
| XSLBWJPPWWFTQY-UHFFFAOYSA-N | $3.5\times10^5$ | | Wang et al. (2017) | Q | 80, 240 |
| MCM:HO1C3OOH | $2.0\times10^4$ | | Wang et al. (2017) | Q | 80, 238 |
| $C_3H_8O_3$ | $2.1\times10^3$ | | Wang et al. (2017) | Q | 80, 239 |
| LTKFKDLZMYUGKU-UHFFFAOYSA-N | $3.6\times10^4$ | | Wang et al. (2017) | Q | 80, 240 |
| MCM:HOC2H4CO3H | $2.6\times10^5$ | | Wang et al. (2017) | Q | 80, 238 |
| $C_3H_6O_4$ | $8.9\times10^4$ | | Wang et al. (2017) | Q | 80, 239 |
| HPIZQAHZLDXVRM-UHFFFAOYSA-N | $1.5\times10^2$ | | Wang et al. (2017) | Q | 80, 240 |
| MCM:HYPROPO2H | $2.6\times10^4$ | | Wang et al. (2017) | Q | 80, 238 |
| $C_3H_8O_3$ | $3.0\times10^3$ | | Wang et al. (2017) | Q | 80, 239 |
| CGHALRGHFXEMJB-UHFFFAOYSA-N | $4.5\times10^3$ | | Wang et al. (2017) | Q | 80, 240 |
| MCM:IPROPOLO2H | $2.6\times10^4$ | | Wang et al. (2017) | Q | 80, 238 |
| $C_3H_8O_3$ | $2.1\times10^3$ | | Wang et al. (2017) | Q | 80, 239 |
| LGTUXDWECPSIQO-UHFFFAOYSA-N | $4.2\times10^3$ | | Wang et al. (2017) | Q | 80, 240 |
| MCM:IPROPOLPER | $1.4\times10^4$ | | Wang et al. (2017) | Q | 80, 238 |
| $C_3H_6O_4$ | $1.5\times10^2$ | | Wang et al. (2017) | Q | 80, 239 |
| LLWVBKQQPPRNEX-UHFFFAOYSA-N | $4.9\times10^1$ | | Wang et al. (2017) | Q | 80, 240 |
| MCM:BUT2OLOOH | $2.5\times10^4$ | | Wang et al. (2017) | Q | 80, 238 |
| $C_4H_{10}O_3$ | $1.5\times10^3$ | | Wang et al. (2017) | Q | 80, 239 |
| VRQPOZSYDWHVQK-UHFFFAOYSA-N | $2.0\times10^3$ | | Wang et al. (2017) | Q | 80, 240 |
| MCM:BUTDAOOH | $6.5\times10^4$ | | Wang et al. (2017) | Q | 80, 238 |
| $C_4H_8O_3$ | $2.5\times10^5$ | | Wang et al. (2017) | Q | 80, 239 |
| ATNKTSDVMCPWQK-UHFFFAOYSA-N | $1.6\times10^5$ | | Wang et al. (2017) | Q | 80, 240 |
| MCM:BUTDBOOH | $5.9\times10^4$ | | Wang et al. (2017) | Q | 80, 238 |
| $C_4H_8O_3$ | $2.3\times10^3$ | | Wang et al. (2017) | Q | 80, 239 |
| VQGAWAMXUXLZCZ-UHFFFAOYSA-N | $2.6\times10^3$ | | Wang et al. (2017) | Q | 80, 240 |
| MCM:BUTDCOOH | $5.9\times10^4$ | | Wang et al. (2017) | Q | 80, 238 |
| $C_4H_8O_3$ | $1.9\times10^3$ | | Wang et al. (2017) | Q | 80, 239 |
| TVUDLONEIXNOQY-UHFFFAOYSA-N | $3.0\times10^3$ | | Wang et al. (2017) | Q | 80, 240 |
| MCM:HC3CCO3H | $3.3\times10^4$ | | Wang et al. (2017) | Q | 80, 238 |
| $C_4H_6O_4$ | $2.2\times10^2$ | | Wang et al. (2017) | Q | 80, 239 |
| ZSVMRAYUVFDTKB-UHFFFAOYSA-N | $1.5\times10^1$ | | Wang et al. (2017) | Q | 80, 240 |



Table A3.4: Peroxides (ROOH) and peroxy radicals (ROO) (...continued)

| Substance Formula (Trivial Name) [CAS Registry Number] InChIKey | $H_s^{cp}$ (at $T^\ominus$) $\left[\dfrac{\text{mol}}{\text{m}^3\,\text{Pa}}\right]$ | $\dfrac{\text{d}\ln H_s^{cp}}{\text{d}(1/T)}$ [K] | Reference | Type | Note |
|---|---|---|---|---|---|
| MCM:HC3CO3H | $9.6\times10^5$ | | Wang et al. (2017) | Q | 80, 238 |
| $C_4H_6O_4$ | $1.9\times10^5$ | | Wang et al. (2017) | Q | 80, 239 |
| AVWVUWGQXKTDKD-UHFFFAOYSA-N | $1.6\times10^3$ | | Wang et al. (2017) | Q | 80, 240 |
| MCM:HMACO3H | $4.9\times10^5$ | | Wang et al. (2017) | Q | 80, 238 |
| $C_4H_6O_4$ | $2.0\times10^4$ | | Wang et al. (2017) | Q | 80, 239 |
| IZGOVIKWSGSSGB-UHFFFAOYSA-N | $9.8\times10^1$ | | Wang et al. (2017) | Q | 80, 240 |
| MCM:HO13C3CO3H | $3.6\times10^7$ | | Wang et al. (2017) | Q | 80, 238 |
| $C_4H_8O_5$ | $9.8\times10^5$ | | Wang et al. (2017) | Q | 80, 239 |
| SAVJJQFYFPVKTL-UHFFFAOYSA-N | $3.6\times10^4$ | | Wang et al. (2017) | Q | 80, 240 |
| MCM:HO13C4OOH | $5.9\times10^7$ | | Wang et al. (2017) | Q | 80, 238 |
| $C_4H_{10}O_4$ | $7.3\times10^6$ | | Wang et al. (2017) | Q | 80, 239 |
| WTIWPVYZJGYFOG-UHFFFAOYSA-N | $1.2\times10^6$ | | Wang et al. (2017) | Q | 80, 240 |
| MCM:HO1C4OOH | $1.6\times10^4$ | | Wang et al. (2017) | Q | 80, 238 |
| $C_4H_{10}O_3$ | $6.9\times10^4$ | | Wang et al. (2017) | Q | 80, 239 |
| RCPPCOTVBKCJGX-UHFFFAOYSA-N | $3.8\times10^3$ | | Wang et al. (2017) | Q | 80, 240 |
| MCM:HO2C3CO3H | $2.5\times10^5$ | | Wang et al. (2017) | Q | 80, 238 |
| $C_4H_8O_4$ | $1.0\times10^5$ | | Wang et al. (2017) | Q | 80, 239 |
| WSXMRWGMMYXEOT-UHFFFAOYSA-N | $4.8\times10^1$ | | Wang et al. (2017) | Q | 80, 240 |
| MCM:HO2C4OOH | $1.9\times10^4$ | | Wang et al. (2017) | Q | 80, 238 |
| $C_4H_{10}O_3$ | $9.1\times10^2$ | | Wang et al. (2017) | Q | 80, 239 |
| VJLLYXNZUMWUFH-UHFFFAOYSA-N | $1.5\times10^4$ | | Wang et al. (2017) | Q | 80, 240 |
| MCM:HO3C3CO3H | $1.2\times10^4$ | | Wang et al. (2017) | Q | 80, 238 |
| $C_4H_8O_4$ | $8.9\times10^1$ | | Wang et al. (2017) | Q | 80, 239 |
| BPXTXQLLUZFFNC-UHFFFAOYSA-N | $1.2\times10^1$ | | Wang et al. (2017) | Q | 80, 240 |
| MCM:HO3C4OOH | $2.1\times10^4$ | | Wang et al. (2017) | Q | 80, 238 |
| $C_4H_{10}O_3$ | $1.4\times10^3$ | | Wang et al. (2017) | Q | 80, 239 |
| RMYCTZOJULYKEL-UHFFFAOYSA-N | $1.5\times10^3$ | | Wang et al. (2017) | Q | 80, 240 |
| MCM:HOC3H6CO3H | $2.3\times10^5$ | | Wang et al. (2017) | Q | 80, 238 |
| $C_4H_8O_4$ | $8.9\times10^4$ | | Wang et al. (2017) | Q | 80, 239 |
| CAGKPIQTEWLESW-UHFFFAOYSA-N | $3.1\times10^2$ | | Wang et al. (2017) | Q | 80, 240 |
| MCM:HOIPRCO3H | $2.5\times10^5$ | | Wang et al. (2017) | Q | 80, 238 |
| $C_4H_8O_4$ | $9.8\times10^4$ | | Wang et al. (2017) | Q | 80, 239 |
| XMJMNOOOSJCSST-UHFFFAOYSA-N | $6.9\times10^1$ | | Wang et al. (2017) | Q | 80, 240 |
| MCM:IBUTOLBO2H | $1.4\times10^4$ | | Wang et al. (2017) | Q | 80, 238 |
| $C_4H_{10}O_3$ | $1.3\times10^3$ | | Wang et al. (2017) | Q | 80, 239 |
| NCECNUIKMRTTTQ-UHFFFAOYSA-N | $1.3\times10^3$ | | Wang et al. (2017) | Q | 80, 240 |
| MCM:IBUTOLCO2H | $1.9\times10^4$ | | Wang et al. (2017) | Q | 80, 238 |
| $C_4H_{10}O_3$ | $1.5\times10^3$ | | Wang et al. (2017) | Q | 80, 239 |
| IYIYQILWOLBTBM-UHFFFAOYSA-N | $1.1\times10^4$ | | Wang et al. (2017) | Q | 80, 240 |





Table A3.4: Peroxides (ROOH) and peroxy radicals (ROO) (...continued)

| Substance<br>Formula<br>(Trivial Name)<br>[CAS Registry Number]<br>InChIKey | $H_s^{cp}$<br>(at $T^{\ominus}$)<br>$\left[\dfrac{\mathrm{mol}}{\mathrm{m}^3\,\mathrm{Pa}}\right]$ | $\dfrac{\mathrm{d}\ln H_s^{cp}}{\mathrm{d}(1/T)}$<br><br>[K] | Reference | Type | Note |
|---|---|---|---|---|---|
| MCM:IPRHOCO3H | $8.1\times10^3$ | | Wang et al. (2017) | Q | 80, 238 |
| $C_4H_8O_4$ | $3.2\times10^1$ | | Wang et al. (2017) | Q | 80, 239 |
| VAHIZEUROSIQMC-UHFFFAOYSA-N | 7.8 | | Wang et al. (2017) | Q | 80, 240 |
| MCM:NBUTOLAOOH | $2.1\times10^4$ | | Wang et al. (2017) | Q | 80, 238 |
| $C_4H_{10}O_3$ | $2.1\times10^3$ | | Wang et al. (2017) | Q | 80, 239 |
| KPUCJKMFRKKADT-UHFFFAOYSA-N | $2.0\times10^3$ | | Wang et al. (2017) | Q | 80, 240 |
| MCM:NBUTOLBOOH | $1.9\times10^4$ | | Wang et al. (2017) | Q | 80, 238 |
| $C_4H_{10}O_3$ | $1.4\times10^3$ | | Wang et al. (2017) | Q | 80, 239 |
| UKXKMFPUVLQYHV-UHFFFAOYSA-N | $2.6\times10^4$ | | Wang et al. (2017) | Q | 80, 240 |
| MCM:TBUTOLOOH | $1.4\times10^4$ | | Wang et al. (2017) | Q | 80, 238 |
| $C_4H_{10}O_3$ | $8.0\times10^2$ | | Wang et al. (2017) | Q | 80, 239 |
| OLMJQPVPWWPTBW-UHFFFAOYSA-N | $2.0\times10^3$ | | Wang et al. (2017) | Q | 80, 240 |
| MCM:C3M3OHCO3H | $1.1\times10^4$ | | Wang et al. (2017) | Q | 80, 238 |
| $C_5H_{10}O_4$ | $7.6\times10^1$ | | Wang et al. (2017) | Q | 80, 239 |
| OPXDDGDNASPZKO-UHFFFAOYSA-N | 8.1 | | Wang et al. (2017) | Q | 80, 240 |
| MCM:C42CO3H | $3.3\times10^7$ | | Wang et al. (2017) | Q | 80, 238 |
| $C_5H_{10}O_5$ | $1.2\times10^6$ | | Wang et al. (2017) | Q | 80, 239 |
| ZMIMRJOKAFWNST-UHFFFAOYSA-N | $1.1\times10^4$ | | Wang et al. (2017) | Q | 80, 240 |
| MCM:C46CO3H | $4.5\times10^5$ | | Wang et al. (2017) | Q | 80, 238 |
| $C_5H_8O_4$ | $1.4\times10^5$ | | Wang et al. (2017) | Q | 80, 239 |
| VWNMYJVYTYSNQP-UHFFFAOYSA-N | $1.3\times10^2$ | | Wang et al. (2017) | Q | 80, 240 |
| MCM:C4OH2CO3H | $9.1\times10^6$ | | Wang et al. (2017) | Q | 80, 238 |
| $C_5H_{10}O_5$ | $2.0\times10^5$ | | Wang et al. (2017) | Q | 80, 239 |
| WQUIEUVQOLAPHC-UHFFFAOYSA-N | $9.6\times10^1$ | | Wang et al. (2017) | Q | 80, 240 |
| MCM:C4OHCO3H | $1.0\times10^4$ | | Wang et al. (2017) | Q | 80, 238 |
| $C_5H_{10}O_4$ | $5.5\times10^1$ | | Wang et al. (2017) | Q | 80, 239 |
| OQVSVCDIVLTQNI-UHFFFAOYSA-N | 6.8 | | Wang et al. (2017) | Q | 80, 240 |
| MCM:C51OH2OOH | $1.9\times10^4$ | | Wang et al. (2017) | Q | 80, 238 |
| $C_5H_{12}O_3$ | $1.7\times10^3$ | | Wang et al. (2017) | Q | 80, 239 |
| YEVLRKKUVOPQLT-UHFFFAOYSA-N | $1.8\times10^3$ | | Wang et al. (2017) | Q | 80, 240 |
| MCM:C524OOH | $1.1\times10^8$ | | Wang et al. (2017) | Q | 80, 238 |
| $C_5H_{10}O_4$ | $1.2\times10^7$ | | Wang et al. (2017) | Q | 80, 239 |
| CMYVSBWTXHBCDK-UHFFFAOYSA-N | $8.5\times10^5$ | | Wang et al. (2017) | Q | 80, 240 |
| MCM:C52OH1OOH | $1.9\times10^4$ | | Wang et al. (2017) | Q | 80, 238 |
| $C_5H_{12}O_3$ | $1.0\times10^3$ | | Wang et al. (2017) | Q | 80, 239 |
| UCQFCLCUMHUGLJ-UHFFFAOYSA-N | $1.0\times10^3$ | | Wang et al. (2017) | Q | 80, 240 |
| MCM:C52OH3OOH | $1.9\times10^4$ | | Wang et al. (2017) | Q | 80, 238 |
| $C_5H_{12}O_3$ | $1.7\times10^3$ | | Wang et al. (2017) | Q | 80, 239 |
| HJJMDLGJAJYCSL-UHFFFAOYSA-N | $1.2\times10^3$ | | Wang et al. (2017) | Q | 80, 240 |



Table A3.4: Peroxides (ROOH) and peroxy radicals (ROO) (...continued)

| Substance Formula (Trivial Name) [CAS Registry Number] InChIKey | $H_s^{cp}$ (at $T^\ominus$) $\left[\dfrac{\mathrm{mol}}{\mathrm{m}^3\,\mathrm{Pa}}\right]$ | $\dfrac{\mathrm{d}\ln H_s^{cp}}{\mathrm{d}(1/T)}$ [K] | Reference | Type | Note |
|---|---|---|---|---|---|
| MCM:C52OOH | $1.5\times10^4$ | | Wang et al. (2017) | Q | 80, 238 |
| $C_5H_{12}O_3$ | $5.1\times10^4$ | | Wang et al. (2017) | Q | 80, 239 |
| ZPHQYWRKWRKGNS-UHFFFAOYSA-N | $3.7\times10^3$ | | Wang et al. (2017) | Q | 80, 240 |
| MCM:C53OH2OOH | $1.9\times10^4$ | | Wang et al. (2017) | Q | 80, 238 |
| $C_5H_{12}O_3$ | $1.0\times10^3$ | | Wang et al. (2017) | Q | 80, 239 |
| AJVYEIIRONFKES-UHFFFAOYSA-N | $8.5\times10^2$ | | Wang et al. (2017) | Q | 80, 240 |
| MCM:C54OOH | $3.2\times10^7$ | | Wang et al. (2017) | Q | 80, 238 |
| $C_5H_{12}O_4$ | $5.9\times10^6$ | | Wang et al. (2017) | Q | 80, 239 |
| YUSGXJKDHCBGQE-UHFFFAOYSA-N | $6.3\times10^5$ | | Wang et al. (2017) | Q | 80, 240 |
| MCM:C56OOH | $5.6\times10^7$ | | Wang et al. (2017) | Q | 80, 238 |
| $C_5H_{12}O_4$ | $7.6\times10^6$ | | Wang et al. (2017) | Q | 80, 239 |
| ZJCLGCDNIIJGCW-UHFFFAOYSA-N | $7.3\times10^5$ | | Wang et al. (2017) | Q | 80, 240 |
| MCM:H13C43CO3H | $1.9\times10^7$ | | Wang et al. (2017) | Q | 80, 238 |
| $C_5H_{10}O_5$ | $4.1\times10^5$ | | Wang et al. (2017) | Q | 80, 239 |
| AHAADFOLDXPWKJ-UHFFFAOYSA-N | $1.5\times10^4$ | | Wang et al. (2017) | Q | 80, 240 |
| MCM:H2M2C3CO3H | $1.4\times10^5$ | | Wang et al. (2017) | Q | 80, 238 |
| $C_5H_{10}O_4$ | $5.9\times10^4$ | | Wang et al. (2017) | Q | 80, 239 |
| WNBRWGLXVYJADJ-UHFFFAOYSA-N | $1.4\times10^1$ | | Wang et al. (2017) | Q | 80, 240 |
| MCM:H2M3C4OOH | $1.7\times10^4$ | | Wang et al. (2017) | Q | 80, 238 |
| $C_5H_{12}O_3$ | $8.5\times10^2$ | | Wang et al. (2017) | Q | 80, 239 |
| ZUPZYIDTDVLJQO-UHFFFAOYSA-N | $6.5\times10^3$ | | Wang et al. (2017) | Q | 80, 240 |
| MCM:HC4ACO3H | $6.5\times10^5$ | | Wang et al. (2017) | Q | 80, 238 |
| $C_5H_8O_4$ | $2.0\times10^5$ | | Wang et al. (2017) | Q | 80, 239 |
| RKJIARBOZVRICG-UHFFFAOYSA-N | $8.3\times10^2$ | | Wang et al. (2017) | Q | 80, 240 |
| MCM:HC4CCO3H | $6.5\times10^5$ | | Wang et al. (2017) | Q | 80, 238 |
| $C_5H_8O_4$ | $1.9\times10^5$ | | Wang et al. (2017) | Q | 80, 239 |
| BKBSBVFBLVOVGK-UHFFFAOYSA-N | $7.4\times10^2$ | | Wang et al. (2017) | Q | 80, 240 |
| MCM:HM22C3OOH | $1.0\times10^4$ | | Wang et al. (2017) | Q | 80, 238 |
| $C_5H_{12}O_3$ | $9.8\times10^2$ | | Wang et al. (2017) | Q | 80, 239 |
| YTVLHPDOOITGNI-UHFFFAOYSA-N | $2.4\times10^3$ | | Wang et al. (2017) | Q | 80, 240 |
| MCM:HM22CO3H | $1.4\times10^5$ | | Wang et al. (2017) | Q | 80, 238 |
| $C_5H_{10}O_4$ | $5.4\times10^4$ | | Wang et al. (2017) | Q | 80, 239 |
| JLDURWTXWMLEQQ-UHFFFAOYSA-N | $1.0\times10^1$ | | Wang et al. (2017) | Q | 80, 240 |
| MCM:HM2C43OOH | $1.7\times10^4$ | | Wang et al. (2017) | Q | 80, 238 |
| $C_5H_{12}O_3$ | $1.6\times10^3$ | | Wang et al. (2017) | Q | 80, 239 |
| HSLVQIVCJSWUOZ-UHFFFAOYSA-N | $8.7\times10^3$ | | Wang et al. (2017) | Q | 80, 240 |
| MCM:HM33C3OOH | $1.0\times10^4$ | | Wang et al. (2017) | Q | 80, 238 |
| $C_5H_{12}O_3$ | $7.3\times10^2$ | | Wang et al. (2017) | Q | 80, 239 |
| WHBWUNCJQHRHKD-UHFFFAOYSA-N | $1.3\times10^4$ | | Wang et al. (2017) | Q | 80, 240 |



Table A3.4: Peroxides (ROOH) and peroxy radicals (ROO) (...continued)

| Substance Formula (Trivial Name) [CAS Registry Number] InChIKey | $H_s^{cp}$ (at $T^\ominus$) $\left[\dfrac{\mathrm{mol}}{\mathrm{m^3\,Pa}}\right]$ | $\dfrac{\mathrm{d}\ln H_s^{cp}}{\mathrm{d}(1/T)}$ [K] | Reference | Type | Note |
|---|---|---|---|---|---|
| MCM:HO13C5OOH | $5.6\times10^7$ | | Wang et al. (2017) | Q | 80, 238 |
| $C_5H_{12}O_4$ | $7.6\times10^6$ | | Wang et al. (2017) | Q | 80, 239 |
| MPQCFHWMMIVYQV-UHFFFAOYSA-N | $1.3\times10^6$ | | Wang et al. (2017) | Q | 80, 240 |
| MCM:HO1C5OOH | $1.5\times10^4$ | | Wang et al. (2017) | Q | 80, 238 |
| $C_5H_{12}O_3$ | $4.8\times10^4$ | | Wang et al. (2017) | Q | 80, 239 |
| NOIFGIIKGVCIDM-UHFFFAOYSA-N | $2.9\times10^3$ | | Wang et al. (2017) | Q | 80, 240 |
| MCM:HO24C4CO3H | $3.3\times10^7$ | | Wang et al. (2017) | Q | 80, 238 |
| $C_5H_{10}O_5$ | $7.6\times10^5$ | | Wang et al. (2017) | Q | 80, 239 |
| BPXJXZFDALRNCD-UHFFFAOYSA-N | $1.8\times10^4$ | | Wang et al. (2017) | Q | 80, 240 |
| MCM:HO24C5OOH | $5.6\times10^7$ | | Wang et al. (2017) | Q | 80, 238 |
| $C_5H_{12}O_4$ | $3.4\times10^6$ | | Wang et al. (2017) | Q | 80, 239 |
| LXCUFKDAZLTFBK-UHFFFAOYSA-N | $7.6\times10^5$ | | Wang et al. (2017) | Q | 80, 240 |
| MCM:HO2C43CO3H | $2.3\times10^5$ | | Wang et al. (2017) | Q | 80, 238 |
| $C_5H_{10}O_4$ | $7.3\times10^4$ | | Wang et al. (2017) | Q | 80, 239 |
| JJSNUFSNLCXQDD-UHFFFAOYSA-N | $3.8\times10^1$ | | Wang et al. (2017) | Q | 80, 240 |
| MCM:HO2C4CO3H | $2.2\times10^5$ | | Wang et al. (2017) | Q | 80, 238 |
| $C_5H_{10}O_4$ | $6.3\times10^4$ | | Wang et al. (2017) | Q | 80, 239 |
| WEYKGJBTAJHRDZ-UHFFFAOYSA-N | $1.1\times10^2$ | | Wang et al. (2017) | Q | 80, 240 |
| MCM:HO2C54OOH | $1.7\times10^4$ | | Wang et al. (2017) | Q | 80, 238 |
| $C_5H_{12}O_3$ | $5.4\times10^2$ | | Wang et al. (2017) | Q | 80, 239 |
| NQEGDBNDYHCERX-UHFFFAOYSA-N | $6.2\times10^3$ | | Wang et al. (2017) | Q | 80, 240 |
| MCM:HO2C5OOH | $7.1\times10^3$ | 11000 | Wieser et al. (2023) | Q | 437 |
| $C_5H_{12}O_3$ | $1.5\times10^4$ | | Wang et al. (2017) | Q | 80, 238 |
| GSRFGQMXWRCOMF-UHFFFAOYSA-N | $5.4\times10^4$ | | Wang et al. (2017) | Q | 80, 239 |
| | $1.4\times10^3$ | | Wang et al. (2017) | Q | 80, 240 |
| MCM:HO2M2C4OOH | $1.0\times10^4$ | | Wang et al. (2017) | Q | 80, 238 |
| $C_5H_{12}O_3$ | $5.4\times10^2$ | | Wang et al. (2017) | Q | 80, 239 |
| BPHNXEOBBWBBGO-UHFFFAOYSA-N | $4.3\times10^3$ | | Wang et al. (2017) | Q | 80, 240 |
| MCM:HO3C4CO3H | $2.2\times10^5$ | | Wang et al. (2017) | Q | 80, 238 |
| $C_5H_{10}O_4$ | $5.6\times10^4$ | | Wang et al. (2017) | Q | 80, 239 |
| OFXAQZFPCOWEPU-UHFFFAOYSA-N | $1.7\times10^1$ | | Wang et al. (2017) | Q | 80, 240 |
| MCM:HO3C5OOH | $1.5\times10^4$ | | Wang et al. (2017) | Q | 80, 238 |
| $C_5H_{12}O_3$ | $6.9\times10^2$ | | Wang et al. (2017) | Q | 80, 239 |
| AMTQAQGRUCFWLQ-UHFFFAOYSA-N | $5.1\times10^3$ | | Wang et al. (2017) | Q | 80, 240 |
| MCM:HOBUT2CO3H | $2.2\times10^5$ | | Wang et al. (2017) | Q | 80, 238 |
| $C_5H_{10}O_4$ | $5.8\times10^4$ | | Wang et al. (2017) | Q | 80, 239 |
| HHXRGEGWJUDFIW-UHFFFAOYSA-N | $1.9\times10^2$ | | Wang et al. (2017) | Q | 80, 240 |
| MCM:HOIBUTCO3H | $2.2\times10^5$ | | Wang et al. (2017) | Q | 80, 238 |
| $C_5H_{10}O_4$ | $5.6\times10^4$ | | Wang et al. (2017) | Q | 80, 239 |
| JVGUWDNKCJEDLO-UHFFFAOYSA-N | $1.4\times10^2$ | | Wang et al. (2017) | Q | 80, 240 |





Table A3.4: Peroxides (ROOH) and peroxy radicals (ROO) (. . . continued)

| Substance Formula (Trivial Name) [CAS Registry Number] InChIKey | $H_s^{cp}$ (at $T^\ominus$) $\left[\dfrac{\text{mol}}{\text{m}^3\,\text{Pa}}\right]$ | $\dfrac{\text{d}\ln H_s^{cp}}{\text{d}(1/T)}$ [K] | Reference | Type | Note |
|---|---|---|---|---|---|
| MCM:ISOPAOOH | $4.4\times10^4$ | | Wang et al. (2017) | Q | 80, 238 |
| $C_5H_{10}O_3$ | $1.7\times10^5$ | | Wang et al. (2017) | Q | 80, 239 |
| PEJOQVDRQFVMLE-UHFFFAOYSA-N | $8.5\times10^4$ | | Wang et al. (2017) | Q | 80, 240 |
| MCM:ISOPBOOH | $1.2\times10^3$ | 9900 | Rivera-Rios (2018) | M | |
| $C_5H_{10}O_3$ | $9.9\times10^2$ | | Rivera-Rios (2018) | Q | 453 |
| QTGGFXPTAQEODO-UHFFFAOYSA-N | $1.3\times10^4$ | | Rivera-Rios (2018) | Q | 454 |
| | $3.3\times10^4$ | | Wang et al. (2017) | Q | 80, 238 |
| | $1.4\times10^3$ | | Wang et al. (2017) | Q | 80, 239 |
| | $4.2\times10^2$ | | Wang et al. (2017) | Q | 80, 240 |
| MCM:ISOPCOOH | $4.4\times10^4$ | | Wang et al. (2017) | Q | 80, 238 |
| $C_5H_{10}O_3$ | $2.0\times10^5$ | | Wang et al. (2017) | Q | 80, 239 |
| FCLDANRCFPUKED-UHFFFAOYSA-N | $1.0\times10^5$ | | Wang et al. (2017) | Q | 80, 240 |
| MCM:ISOPDOOH | $1.2\times10^2$ | 7500 | Rivera-Rios (2018) | M | |
| $C_5H_{10}O_3$ | $2.6\times10^4$ | | Rivera-Rios (2018) | Q | 453 |
| HYHMYONEYJINON-UHFFFAOYSA-N | $1.3\times10^4$ | | Rivera-Rios (2018) | Q | 454 |
| | $4.0\times10^4$ | | Wang et al. (2017) | Q | 80, 238 |
| | $1.8\times10^3$ | | Wang et al. (2017) | Q | 80, 239 |
| | $8.1\times10^2$ | | Wang et al. (2017) | Q | 80, 240 |
| MCM:M2BU2OLOOH | $1.4\times10^4$ | | Wang et al. (2017) | Q | 80, 238 |
| $C_5H_{12}O_3$ | $1.1\times10^3$ | | Wang et al. (2017) | Q | 80, 239 |
| YWZMEYHQKKHRJF-UHFFFAOYSA-N | $2.1\times10^3$ | | Wang et al. (2017) | Q | 80, 240 |
| MCM:M2BUOL2OOH | $1.2\times10^4$ | | Wang et al. (2017) | Q | 80, 238 |
| $C_5H_{12}O_3$ | $1.1\times10^3$ | | Wang et al. (2017) | Q | 80, 239 |
| KLOVYRXVQOACLZ-UHFFFAOYSA-N | $6.6\times10^2$ | | Wang et al. (2017) | Q | 80, 240 |
| MCM:M3BU2OLOOH | $1.9\times10^4$ | | Wang et al. (2017) | Q | 80, 238 |
| $C_5H_{12}O_3$ | $1.1\times10^3$ | | Wang et al. (2017) | Q | 80, 239 |
| AFEDOYRLNKBWPZ-UHFFFAOYSA-N | $8.7\times10^2$ | | Wang et al. (2017) | Q | 80, 240 |
| MCM:MBOAOOH | $4.1\times10^7$ | | Wang et al. (2017) | Q | 80, 238 |
| $C_5H_{12}O_4$ | $4.5\times10^6$ | | Wang et al. (2017) | Q | 80, 239 |
| VJEOLZUQMDEVMO-UHFFFAOYSA-N | $6.8\times10^3$ | | Wang et al. (2017) | Q | 80, 240 |
| MCM:MBOBOOH | $1.4\times10^7$ | | Wang et al. (2017) | Q | 80, 238 |
| $C_5H_{12}O_4$ | $6.3\times10^5$ | | Wang et al. (2017) | Q | 80, 239 |
| GXJWPCHTRDIVHR-UHFFFAOYSA-N | $5.1\times10^3$ | | Wang et al. (2017) | Q | 80, 240 |
| MCM:ME2BUOLOOH | $1.4\times10^4$ | | Wang et al. (2017) | Q | 80, 238 |
| $C_5H_{12}O_3$ | $6.9\times10^2$ | | Wang et al. (2017) | Q | 80, 239 |
| SAXHQAUHEBEDAO-UHFFFAOYSA-N | $9.3\times10^2$ | | Wang et al. (2017) | Q | 80, 240 |
| MCM:ME3BUOLOOH | $1.9\times10^4$ | | Wang et al. (2017) | Q | 80, 238 |
| $C_5H_{12}O_3$ | $2.3\times10^3$ | | Wang et al. (2017) | Q | 80, 239 |
| MUYQYONVNPPJGI-UHFFFAOYSA-N | $9.8\times10^2$ | | Wang et al. (2017) | Q | 80, 240 |



Table A3.4: Peroxides (ROOH) and peroxy radicals (ROO) (...continued)

| Substance Formula (Trivial Name) [CAS Registry Number] InChIKey | $H_s^{cp}$ (at $T^{\ominus}$) $\left[\dfrac{\text{mol}}{\text{m}^3\,\text{Pa}}\right]$ | $\dfrac{\text{d}\ln H_s^{cp}}{\text{d}(1/T)}$ [K] | Reference | Type | Note |
|---|---|---|---|---|---|
| MCM:PROL11MOOH | $1.2\times10^4$ | | Wang et al. (2017) | Q | 80, 238 |
| $C_5H_{12}O_3$ | $5.6\times10^2$ | | Wang et al. (2017) | Q | 80, 239 |
| UVDGWSPEOVCOFS-UHFFFAOYSA-N | $1.5\times10^3$ | | Wang et al. (2017) | Q | 80, 240 |
| MCM:PROL1MCO3H | $6.3\times10^3$ | | Wang et al. (2017) | Q | 80, 238 |
| $C_5H_{10}O_4$ | $2.1\times10^1$ | | Wang et al. (2017) | Q | 80, 239 |
| KTQROZYFUGXCGA-UHFFFAOYSA-N | 2.2 | | Wang et al. (2017) | Q | 80, 240 |
| MCM:BZBIPER2OH | $3.0\times10^7$ | | Wang et al. (2017) | Q | 80, 238 |
| $C_6H_8O_4$ | $1.9\times10^6$ | | Wang et al. (2017) | Q | 80, 239 |
| BVNIDXNYDXJAFS-UHFFFAOYSA-N | $8.5\times10^6$ | | Wang et al. (2017) | Q | 80, 240 |
| MCM:BZBIPEROOH | $7.1\times10^8$ | | Wang et al. (2017) | Q | 80, 238 |
| $C_6H_8O_5$ | $9.3\times10^6$ | | Wang et al. (2017) | Q | 80, 239 |
| MXPMOVDECINDPN-UHFFFAOYSA-N | $2.3\times10^6$ | | Wang et al. (2017) | Q | 80, 240 |
| MCM:C4ME2OHOOH | $7.6\times10^3$ | | Wang et al. (2017) | Q | 80, 238 |
| $C_6H_{14}O_3$ | $3.8\times10^2$ | | Wang et al. (2017) | Q | 80, 239 |
| HOGFDWYRBLIMGM-UHFFFAOYSA-N | $1.0\times10^3$ | | Wang et al. (2017) | Q | 80, 240 |
| MCM:C518CO3H | $4.3\times10^5$ | | Wang et al. (2017) | Q | 80, 238 |
| $C_6H_{10}O_4$ | $5.6\times10^4$ | | Wang et al. (2017) | Q | 80, 239 |
| HLKDNTMYLJJNIF-UHFFFAOYSA-N | $5.4\times10^1$ | | Wang et al. (2017) | Q | 80, 240 |
| MCM:C622OOH | $1.3\times10^4$ | 12000 | Wieser et al. (2023) | Q | 437 |
| $C_6H_{12}O_3$ | $2.9\times10^4$ | | Wang et al. (2017) | Q | 80, 238 |
| GWTGWFHICNOSTQ-UHFFFAOYSA-N | $1.3\times10^3$ | | Wang et al. (2017) | Q | 80, 239 |
| | $1.6\times10^3$ | | Wang et al. (2017) | Q | 80, 240 |
| MCM:C624OOH | $2.9\times10^4$ | | Wang et al. (2017) | Q | 80, 238 |
| $C_6H_{12}O_3$ | $1.3\times10^3$ | | Wang et al. (2017) | Q | 80, 239 |
| DDEGWHJCSBWJSM-UHFFFAOYSA-N | $8.0\times10^3$ | | Wang et al. (2017) | Q | 80, 240 |
| MCM:C64OH5OOH | $1.8\times10^4$ | | Wang et al. (2017) | Q | 80, 238 |
| $C_6H_{14}O_3$ | $7.8\times10^2$ | | Wang et al. (2017) | Q | 80, 239 |
| ZEOCDIHFKNFYIT-UHFFFAOYSA-N | $6.3\times10^2$ | | Wang et al. (2017) | Q | 80, 240 |
| MCM:C65OH4OOH | $1.8\times10^4$ | | Wang et al. (2017) | Q | 80, 238 |
| $C_6H_{14}O_3$ | $1.4\times10^3$ | | Wang et al. (2017) | Q | 80, 239 |
| PVVIDEIOKNUTFV-UHFFFAOYSA-N | $1.0\times10^3$ | | Wang et al. (2017) | Q | 80, 240 |
| MCM:C6OH5OOH | $1.5\times10^4$ | | Wang et al. (2017) | Q | 80, 238 |
| $C_6H_{14}O_3$ | $1.3\times10^3$ | | Wang et al. (2017) | Q | 80, 239 |
| HRDCBAWVKIWIOZ-UHFFFAOYSA-N | $2.0\times10^3$ | | Wang et al. (2017) | Q | 80, 240 |
| MCM:CYHXOLAOOH | $5.0\times10^4$ | | Wang et al. (2017) | Q | 80, 238 |
| $C_6H_{12}O_3$ | $8.5\times10^4$ | | Wang et al. (2017) | Q | 80, 239 |
| UBIBQRDHLFTCKX-UHFFFAOYSA-N | $6.5\times10^3$ | | Wang et al. (2017) | Q | 80, 240 |
| MCM:H13M3C5OOH | $3.0\times10^7$ | | Wang et al. (2017) | Q | 80, 238 |
| $C_6H_{14}O_4$ | $7.3\times10^6$ | | Wang et al. (2017) | Q | 80, 239 |
| KEIPMCKLLYLETP-UHFFFAOYSA-N | $1.0\times10^6$ | | Wang et al. (2017) | Q | 80, 240 |





Table A3.4: Peroxides (ROOH) and peroxy radicals (ROO) (...continued)

| Substance Formula (Trivial Name) [CAS Registry Number] InChIKey | $H_s^{cp}$ (at $T^{\ominus}$) $\left[\dfrac{\mathrm{mol}}{\mathrm{m^3\,Pa}}\right]$ | $\dfrac{\mathrm{d}\ln H_s^{cp}}{\mathrm{d}(1/T)}$ [K] | Reference | Type | Note |
|---|---|---|---|---|---|
| MCM:H1MC5OOH | $1.4\times10^4$ | | Wang et al. (2017) | Q | 80, 238 |
| $C_6H_{14}O_3$ | $5.5\times10^4$ | | Wang et al. (2017) | Q | 80, 239 |
| NRUVZELSEXXOEO-UHFFFAOYSA-N | $2.8\times10^3$ | | Wang et al. (2017) | Q | 80, 240 |
| MCM:H2M2C4CO3H | $1.3\times10^5$ | | Wang et al. (2017) | Q | 80, 238 |
| $C_6H_{12}O_4$ | $4.6\times10^4$ | | Wang et al. (2017) | Q | 80, 239 |
| NRXKHRBCIJLLIW-UHFFFAOYSA-N | $1.2\times10^2$ | | Wang et al. (2017) | Q | 80, 240 |
| MCM:H2M3C4CO3H | $2.0\times10^5$ | | Wang et al. (2017) | Q | 80, 238 |
| $C_6H_{12}O_4$ | $5.0\times10^4$ | | Wang et al. (2017) | Q | 80, 239 |
| BSSSRXSGDLZQTD-UHFFFAOYSA-N | $6.2\times10^1$ | | Wang et al. (2017) | Q | 80, 240 |
| MCM:H2MC5OOH | $1.4\times10^4$ | | Wang et al. (2017) | Q | 80, 238 |
| $C_6H_{14}O_3$ | $4.9\times10^4$ | | Wang et al. (2017) | Q | 80, 239 |
| VFABJBQUVDJTMS-UHFFFAOYSA-N | $1.3\times10^3$ | | Wang et al. (2017) | Q | 80, 240 |
| MCM:H3M2C4CO3H | $2.0\times10^5$ | | Wang et al. (2017) | Q | 80, 238 |
| $C_6H_{12}O_4$ | $4.6\times10^4$ | | Wang et al. (2017) | Q | 80, 239 |
| BXUACBWFNFVFRM-UHFFFAOYSA-N | $8.0$ | | Wang et al. (2017) | Q | 80, 240 |
| MCM:H3M3C4CO3H | $1.3\times10^5$ | | Wang et al. (2017) | Q | 80, 238 |
| $C_6H_{12}O_4$ | $4.1\times10^4$ | | Wang et al. (2017) | Q | 80, 239 |
| QREFYDOYYWEEBY-UHFFFAOYSA-N | $5.9$ | | Wang et al. (2017) | Q | 80, 240 |
| MCM:H3M3C5OOH | $8.1\times10^3$ | | Wang et al. (2017) | Q | 80, 238 |
| $C_6H_{14}O_3$ | $4.6\times10^2$ | | Wang et al. (2017) | Q | 80, 239 |
| SFWIETIRFCEPBV-UHFFFAOYSA-N | $2.2\times10^3$ | | Wang et al. (2017) | Q | 80, 240 |
| MCM:HM22C3CO3H | $1.3\times10^5$ | | Wang et al. (2017) | Q | 80, 238 |
| $C_6H_{12}O_4$ | $3.5\times10^4$ | | Wang et al. (2017) | Q | 80, 239 |
| CFTUKLJUHGJIMT-UHFFFAOYSA-N | $2.0\times10^1$ | | Wang et al. (2017) | Q | 80, 240 |
| MCM:HM22C4OOH | $8.1\times10^3$ | | Wang et al. (2017) | Q | 80, 238 |
| $C_6H_{14}O_3$ | $3.2\times10^4$ | | Wang et al. (2017) | Q | 80, 239 |
| RBKQZGUZSABUAQ-UHFFFAOYSA-N | $1.3\times10^3$ | | Wang et al. (2017) | Q | 80, 240 |
| MCM:HM23C4OOH | $1.4\times10^4$ | | Wang et al. (2017) | Q | 80, 238 |
| $C_6H_{14}O_3$ | $6.9\times10^4$ | | Wang et al. (2017) | Q | 80, 239 |
| OEDZXCCMDSIVQE-UHFFFAOYSA-N | $1.9\times10^3$ | | Wang et al. (2017) | Q | 80, 240 |
| MCM:HM2C43CO3H | $2.0\times10^5$ | | Wang et al. (2017) | Q | 80, 238 |
| $C_6H_{12}O_4$ | $5.5\times10^4$ | | Wang et al. (2017) | Q | 80, 239 |
| AAGQWOGOWVBCOC-UHFFFAOYSA-N | $1.1\times10^2$ | | Wang et al. (2017) | Q | 80, 240 |
| MCM:HM33C3CO3H | $1.3\times10^5$ | | Wang et al. (2017) | Q | 80, 238 |
| $C_6H_{12}O_4$ | $3.8\times10^4$ | | Wang et al. (2017) | Q | 80, 239 |
| CJLFHOYFCBYSAF-UHFFFAOYSA-N | $1.0\times10^2$ | | Wang et al. (2017) | Q | 80, 240 |
| MCM:HM33C4OOH | $8.1\times10^3$ | | Wang et al. (2017) | Q | 80, 238 |
| $C_6H_{14}O_3$ | $4.5\times10^4$ | | Wang et al. (2017) | Q | 80, 239 |
| IRDKGYBPQHNNSI-UHFFFAOYSA-N | $6.0\times10^3$ | | Wang et al. (2017) | Q | 80, 240 |





Table A3.4: Peroxides (ROOH) and peroxy radicals (ROO) (…continued)

| Substance Formula (Trivial Name) [CAS Registry Number] InChIKey | $H_s^{cp}$ (at $T^{\ominus}$) $\left[\dfrac{\text{mol}}{\text{m}^3\,\text{Pa}}\right]$ | $\dfrac{\text{d}\ln H_s^{cp}}{\text{d}(1/T)}$ [K] | Reference | Type | Note |
|---|---|---|---|---|---|
| MCM:HO1C6OOH $C_6H_{14}O_3$ AODLLEVTDATZKZ-UHFFFAOYSA-N | $1.2\times10^4$ $3.7\times10^4$ $1.5\times10^3$ | | Wang et al. (2017) Wang et al. (2017) Wang et al. (2017) | Q Q Q | 80, 238 80, 239 80, 240 |
| MCM:HO1MC5OOH $C_6H_{14}O_3$ XFJAYUAZOYAZNW-UHFFFAOYSA-N | $1.4\times10^4$ $2.5\times10^4$ $2.6\times10^3$ | | Wang et al. (2017) Wang et al. (2017) Wang et al. (2017) | Q Q Q | 80, 238 80, 239 80, 240 |
| MCM:HO2C54CO3H $C_6H_{12}O_4$ HFXNQHIGYFOTHL-UHFFFAOYSA-N | $2.0\times10^5$ $4.3\times10^4$ $1.4\times10^2$ | | Wang et al. (2017) Wang et al. (2017) Wang et al. (2017) | Q Q Q | 80, 238 80, 239 80, 240 |
| MCM:HO2C6OOH $C_6H_{14}O_3$ BVOYWPYSMMRYPX-UHFFFAOYSA-N | $5.4\times10^3$ $1.4\times10^4$ $2.4\times10^4$ $1.8\times10^3$ | 11000 | Wieser et al. (2023) Wang et al. (2017) Wang et al. (2017) Wang et al. (2017) | Q Q Q Q | 437 80, 238 80, 239 80, 240 |
| MCM:HO2M2C5OOH $C_6H_{14}O_3$ XDJWSJNLVSIJHD-UHFFFAOYSA-N | $8.1\times10^3$ $4.1\times10^4$ $5.9\times10^2$ | | Wang et al. (2017) Wang et al. (2017) Wang et al. (2017) | Q Q Q | 80, 238 80, 239 80, 240 |
| MCM:HO2MC5OOH $C_6H_{14}O_3$ AGNBGPUCVJMPGZ-UHFFFAOYSA-N | $1.4\times10^4$ $4.8\times10^4$ $4.1\times10^2$ | | Wang et al. (2017) Wang et al. (2017) Wang et al. (2017) | Q Q Q | 80, 238 80, 239 80, 240 |
| MCM:HO3C5CO3H $C_6H_{12}O_4$ NIDIDJCFEJYKBD-UHFFFAOYSA-N | $1.8\times10^5$ $4.5\times10^4$ $4.9\times10^1$ | | Wang et al. (2017) Wang et al. (2017) Wang et al. (2017) | Q Q Q | 80, 238 80, 239 80, 240 |
| MCM:HO3C6OOH $C_6H_{14}O_3$ WSWZWPBQPMRENX-UHFFFAOYSA-N | $1.2\times10^4$ $4.0\times10^4$ $8.0\times10^2$ | | Wang et al. (2017) Wang et al. (2017) Wang et al. (2017) | Q Q Q | 80, 238 80, 239 80, 240 |
| MCM:HO4C5CO3H $C_6H_{12}O_4$ YRMQNGAQYQCFSB-UHFFFAOYSA-N | $1.8\times10^5$ $4.2\times10^4$ $1.3\times10^1$ | | Wang et al. (2017) Wang et al. (2017) Wang et al. (2017) | Q Q Q | 80, 238 80, 239 80, 240 |
| MCM:HO5C5CO3H $C_6H_{12}O_4$ ZOLITBGWKBFVQG-UHFFFAOYSA-N | $8.5\times10^3$ $4.0\times10^1$ $5.9$ | | Wang et al. (2017) Wang et al. (2017) Wang et al. (2017) | Q Q Q | 80, 238 80, 239 80, 240 |
| MCM:HO5C6OOH $C_6H_{14}O_3$ VKWHJFIJUJODAG-UHFFFAOYSA-N | $1.5\times10^4$ $7.6\times10^2$ $1.1\times10^3$ | | Wang et al. (2017) Wang et al. (2017) Wang et al. (2017) | Q Q Q | 80, 238 80, 239 80, 240 |
| MCM:PHENOH $C_6H_8O_5$ JRROYJLQPIXCRD-UHFFFAOYSA-N | $5.5\times10^{10}$ $4.5\times10^8$ $1.2\times10^8$ | | Wang et al. (2017) Wang et al. (2017) Wang et al. (2017) | Q Q Q | 80, 238 80, 239 80, 240 |
| MCM:PHENOOH $C_6H_8O_6$ WQYYYMQUZFKHHO-UHFFFAOYSA-N | $1.4\times10^{12}$ $2.0\times10^8$ $1.4\times10^6$ | | Wang et al. (2017) Wang et al. (2017) Wang et al. (2017) | Q Q Q | 80, 238 80, 239 80, 240 |





Table A3.4: Peroxides (ROOH) and peroxy radicals (ROO) (...continued)

| Substance Formula (Trivial Name) [CAS Registry Number] InChIKey | $H_s^{cp}$ (at $T^\ominus$) $\left[\dfrac{\mathrm{mol}}{\mathrm{m^3\,Pa}}\right]$ | $\dfrac{\mathrm{d}\ln H_s^{cp}}{\mathrm{d}(1/T)}$ [K] | Reference | Type | Note |
|---|---|---|---|---|---|
| MCM:C622CO3H | $3.3\times10^5$ | | Wang et al. (2017) | Q | 80, 238 |
| $C_7H_{12}O_4$ | $3.5\times10^4$ | | Wang et al. (2017) | Q | 80, 239 |
| JSEAPEOJJJOJEL-UHFFFAOYSA-N | $9.3\times10^1$ | | Wang et al. (2017) | Q | 80, 240 |
| MCM:C624CO3H | $3.3\times10^5$ | | Wang et al. (2017) | Q | 80, 238 |
| $C_7H_{12}O_4$ | $3.9\times10^4$ | | Wang et al. (2017) | Q | 80, 239 |
| VELNAHZLHYPPOS-UHFFFAOYSA-N | $2.4\times10^2$ | | Wang et al. (2017) | Q | 80, 240 |
| MCM:C720OOH | $1.2\times10^5$ | | Wang et al. (2017) | Q | 80, 238 |
| $C_7H_{12}O_3$ | $2.9\times10^5$ | | Wang et al. (2017) | Q | 80, 239 |
| LPMHECUYQWFTDP-UHFFFAOYSA-N | $2.4\times10^5$ | | Wang et al. (2017) | Q | 80, 240 |
| MCM:CRESOH | $3.0\times10^{10}$ | | Wang et al. (2017) | Q | 80, 238 |
| $C_7H_{10}O_5$ | $1.7\times10^8$ | | Wang et al. (2017) | Q | 80, 239 |
| UMHUAUCNJCQCRC-UHFFFAOYSA-N | $1.6\times10^9$ | | Wang et al. (2017) | Q | 80, 240 |
| MCM:CRESOOH | $7.8\times10^{11}$ | | Wang et al. (2017) | Q | 80, 238 |
| $C_7H_{10}O_6$ | $7.6\times10^7$ | | Wang et al. (2017) | Q | 80, 239 |
| POYQJFWARUGMBT-UHFFFAOYSA-N | $2.5\times10^6$ | | Wang et al. (2017) | Q | 80, 240 |
| MCM:H2M2C65OOH | $7.6\times10^3$ | | Wang et al. (2017) | Q | 80, 238 |
| $C_7H_{16}O_3$ | $2.9\times10^4$ | | Wang et al. (2017) | Q | 80, 239 |
| MDRLNEUVCFZYIZ-UHFFFAOYSA-N | $1.0\times10^3$ | | Wang et al. (2017) | Q | 80, 240 |
| MCM:H2M4C65OOH | $1.3\times10^4$ | | Wang et al. (2017) | Q | 80, 238 |
| $C_7H_{16}O_3$ | $3.9\times10^4$ | | Wang et al. (2017) | Q | 80, 239 |
| HGFYTCXFDJYCQF-UHFFFAOYSA-N | $1.2\times10^3$ | | Wang et al. (2017) | Q | 80, 240 |
| MCM:H2M5C65OOH | $7.6\times10^3$ | | Wang et al. (2017) | Q | 80, 238 |
| $C_7H_{16}O_3$ | $1.0\times10^4$ | | Wang et al. (2017) | Q | 80, 239 |
| NIKXNAUPHRWTEG-UHFFFAOYSA-N | $2.3\times10^3$ | | Wang et al. (2017) | Q | 80, 240 |
| MCM:H3M3C5CO3H | $9.8\times10^4$ | | Wang et al. (2017) | Q | 80, 238 |
| $C_7H_{14}O_4$ | $3.8\times10^4$ | | Wang et al. (2017) | Q | 80, 239 |
| DTTYYTHTMWEWDN-UHFFFAOYSA-N | $4.8\times10^1$ | | Wang et al. (2017) | Q | 80, 240 |
| MCM:H3M3C6OOH | $6.6\times10^3$ | | Wang et al. (2017) | Q | 80, 238 |
| $C_7H_{16}O_3$ | $3.3\times10^4$ | | Wang et al. (2017) | Q | 80, 239 |
| IWLJNMJTDMGLQI-UHFFFAOYSA-N | $3.6\times10^2$ | | Wang et al. (2017) | Q | 80, 240 |
| MCM:HO3C76OOH | $1.1\times10^4$ | | Wang et al. (2017) | Q | 80, 238 |
| $C_7H_{16}O_3$ | $2.0\times10^4$ | | Wang et al. (2017) | Q | 80, 239 |
| YDKPOQVDDHNRNK-UHFFFAOYSA-N | $9.8\times10^2$ | | Wang et al. (2017) | Q | 80, 240 |
| MCM:HO5C6CO3H | $1.4\times10^5$ | | Wang et al. (2017) | Q | 80, 238 |
| $C_7H_{14}O_4$ | $3.3\times10^4$ | | Wang et al. (2017) | Q | 80, 239 |
| KSOZMYOJLUVTBJ-UHFFFAOYSA-N | $1.0\times10^1$ | | Wang et al. (2017) | Q | 80, 240 |
| MCM:HO6C7OOH | $1.2\times10^4$ | | Wang et al. (2017) | Q | 80, 238 |
| $C_7H_{16}O_3$ | $5.9\times10^2$ | | Wang et al. (2017) | Q | 80, 239 |
| YGTQIECBHSKCOV-UHFFFAOYSA-N | $1.1\times10^3$ | | Wang et al. (2017) | Q | 80, 240 |





Table A3.4: Peroxides (ROOH) and peroxy radicals (ROO) (...continued)

| Substance<br>Formula<br>(Trivial Name)<br>[CAS Registry Number]<br>InChIKey | $H_s^{cp}$<br>(at $T^\ominus$)<br>$\left[\dfrac{\text{mol}}{\text{m}^3\,\text{Pa}}\right]$ | $\dfrac{\text{d}\ln H_s^{cp}}{\text{d}(1/T)}$<br><br>[K] | Reference | Type | Note |
|---|---|---|---|---|---|
| MCM:TLBIPER2OH | $1.7\times10^7$ | | Wang et al. (2017) | Q | 80, 238 |
| $C_7H_{10}O_4$ | $9.6\times10^5$ | | Wang et al. (2017) | Q | 80, 239 |
| CXEVRKZODGVABW-UHFFFAOYSA-N | $2.6\times10^6$ | | Wang et al. (2017) | Q | 80, 240 |
| MCM:TLBIPEROOH | $3.9\times10^8$ | | Wang et al. (2017) | Q | 80, 238 |
| $C_7H_{10}O_5$ | $3.8\times10^6$ | | Wang et al. (2017) | Q | 80, 239 |
| JAKOWCUSWIMTAF-UHFFFAOYSA-N | $9.8\times10^6$ | | Wang et al. (2017) | Q | 80, 240 |
| MCM:EBENZOLOH | $2.5\times10^{10}$ | | Wang et al. (2017) | Q | 80, 238 |
| $C_8H_{12}O_5$ | $1.0\times10^8$ | | Wang et al. (2017) | Q | 80, 239 |
| BUWSXDLGJSAPEF-UHFFFAOYSA-N | $6.0\times10^7$ | | Wang et al. (2017) | Q | 80, 240 |
| MCM:EBENZOLOOH | $6.3\times10^{11}$ | | Wang et al. (2017) | Q | 80, 238 |
| $C_8H_{12}O_6$ | $4.8\times10^7$ | | Wang et al. (2017) | Q | 80, 239 |
| RHHRSXIXEXUGSZ-UHFFFAOYSA-N | $3.9\times10^6$ | | Wang et al. (2017) | Q | 80, 240 |
| MCM:EBZBPER2OH | $1.4\times10^7$ | | Wang et al. (2017) | Q | 80, 238 |
| $C_8H_{12}O_4$ | $6.3\times10^5$ | | Wang et al. (2017) | Q | 80, 239 |
| AUMASFBINMOOAH-UHFFFAOYSA-N | $3.8\times10^6$ | | Wang et al. (2017) | Q | 80, 240 |
| MCM:EBZBPEROOH | $3.2\times10^8$ | | Wang et al. (2017) | Q | 80, 238 |
| $C_8H_{12}O_5$ | $2.5\times10^6$ | | Wang et al. (2017) | Q | 80, 239 |
| SCDRKAWXSVOIOT-UHFFFAOYSA-N | $1.7\times10^6$ | | Wang et al. (2017) | Q | 80, 240 |
| MCM:HO3C86OOH | $1.0\times10^4$ | | Wang et al. (2017) | Q | 80, 238 |
| $C_8H_{18}O_3$ | $1.4\times10^4$ | | Wang et al. (2017) | Q | 80, 239 |
| SPJZDLLIJATZHB-UHFFFAOYSA-N | $2.8\times10^2$ | | Wang et al. (2017) | Q | 80, 240 |
| MCM:HO6C7CO3H | $1.3\times10^5$ | | Wang et al. (2017) | Q | 80, 238 |
| $C_8H_{16}O_4$ | $2.9\times10^4$ | | Wang et al. (2017) | Q | 80, 239 |
| UKHVIUJOBBSNIJ-UHFFFAOYSA-N | $9.1$ | | Wang et al. (2017) | Q | 80, 240 |
| MCM:HO7C8OOH | $1.1\times10^4$ | | Wang et al. (2017) | Q | 80, 238 |
| $C_8H_{18}O_3$ | $4.7\times10^2$ | | Wang et al. (2017) | Q | 80, 239 |
| NDXYTTLTSMWWGD-UHFFFAOYSA-N | $7.4\times10^2$ | | Wang et al. (2017) | Q | 80, 240 |
| MCM:MXYBPER2OH | $9.3\times10^6$ | | Wang et al. (2017) | Q | 80, 238 |
| $C_8H_{12}O_4$ | $4.0\times10^5$ | | Wang et al. (2017) | Q | 80, 239 |
| HJWPGRMMUKJUOW-UHFFFAOYSA-N | $1.5\times10^6$ | | Wang et al. (2017) | Q | 80, 240 |
| MCM:MXYBPEROOH | $2.2\times10^8$ | | Wang et al. (2017) | Q | 80, 238 |
| $C_8H_{12}O_5$ | $1.5\times10^6$ | | Wang et al. (2017) | Q | 80, 239 |
| JLBHNTWNBASURB-UHFFFAOYSA-N | $1.2\times10^6$ | | Wang et al. (2017) | Q | 80, 240 |
| MCM:MXYOLOH | $1.7\times10^{10}$ | | Wang et al. (2017) | Q | 80, 238 |
| $C_8H_{12}O_5$ | $7.4\times10^7$ | | Wang et al. (2017) | Q | 80, 239 |
| HYWULKMMJLAVGS-UHFFFAOYSA-N | $1.5\times10^7$ | | Wang et al. (2017) | Q | 80, 240 |
| MCM:MXYOLOOH | $4.4\times10^{11}$ | | Wang et al. (2017) | Q | 80, 238 |
| $C_8H_{12}O_6$ | $3.2\times10^7$ | | Wang et al. (2017) | Q | 80, 239 |
| GRFXDNYNQVBVKX-UHFFFAOYSA-N | $3.2\times10^7$ | | Wang et al. (2017) | Q | 80, 240 |



Table A3.4: Peroxides (ROOH) and peroxy radicals (ROO) (...continued)

| Substance Formula (Trivial Name) [CAS Registry Number] InChIKey | $H_s^{cp}$ (at $T^\ominus$) $\left[\dfrac{\mathrm{mol}}{\mathrm{m^3\,Pa}}\right]$ | $\dfrac{\mathrm{d}\ln H_s^{cp}}{\mathrm{d}(1/T)}$ [K] | Reference | Type | Note |
|---|---|---|---|---|---|
| MCM:OXYBPER2OH | $9.3\times10^6$ | | Wang et al. (2017) | Q | 80, 238 |
| $C_8H_{12}O_4$ | $5.0\times10^5$ | | Wang et al. (2017) | Q | 80, 239 |
| VCLGCGXZSIRZEA-UHFFFAOYSA-N | $4.2\times10^4$ | | Wang et al. (2017) | Q | 80, 240 |
| MCM:OXYBPEROOH | $2.2\times10^8$ | | Wang et al. (2017) | Q | 80, 238 |
| $C_8H_{12}O_5$ | $1.5\times10^6$ | | Wang et al. (2017) | Q | 80, 239 |
| AMPOFIFBYJNSOH-UHFFFAOYSA-N | $2.0\times10^5$ | | Wang et al. (2017) | Q | 80, 240 |
| MCM:OXYOLOH | $2.0\times10^{10}$ | | Wang et al. (2017) | Q | 80, 238 |
| $C_8H_{12}O_5$ | $1.2\times10^8$ | | Wang et al. (2017) | Q | 80, 239 |
| CAIZPBLFMULAII-UHFFFAOYSA-N | $4.0\times10^7$ | | Wang et al. (2017) | Q | 80, 240 |
| MCM:OXYOLOOH | $5.3\times10^{11}$ | | Wang et al. (2017) | Q | 80, 238 |
| $C_8H_{12}O_6$ | $5.5\times10^7$ | | Wang et al. (2017) | Q | 80, 239 |
| QYYRJAPHRBSJOE-UHFFFAOYSA-N | $8.0\times10^5$ | | Wang et al. (2017) | Q | 80, 240 |
| MCM:PXYBPER2OH | $9.3\times10^6$ | | Wang et al. (2017) | Q | 80, 238 |
| $C_8H_{12}O_4$ | $5.0\times10^5$ | | Wang et al. (2017) | Q | 80, 239 |
| IOEDFCIBQPWSMP-UHFFFAOYSA-N | $4.7\times10^4$ | | Wang et al. (2017) | Q | 80, 240 |
| MCM:PXYBPEROOH | $2.2\times10^8$ | | Wang et al. (2017) | Q | 80, 238 |
| $C_8H_{12}O_5$ | $1.6\times10^6$ | | Wang et al. (2017) | Q | 80, 239 |
| LZVOCZWDZPUJRQ-UHFFFAOYSA-N | $5.1\times10^5$ | | Wang et al. (2017) | Q | 80, 240 |
| MCM:PXYOLOH | $2.0\times10^{10}$ | | Wang et al. (2017) | Q | 80, 238 |
| $C_8H_{12}O_5$ | $1.6\times10^8$ | | Wang et al. (2017) | Q | 80, 239 |
| YSBIHFSZRPVTOU-UHFFFAOYSA-N | $9.6\times10^6$ | | Wang et al. (2017) | Q | 80, 240 |
| MCM:PXYOLOOH | $5.3\times10^{11}$ | | Wang et al. (2017) | Q | 80, 238 |
| $C_8H_{12}O_6$ | $6.8\times10^7$ | | Wang et al. (2017) | Q | 80, 239 |
| ROOFTVJQSDLVOJ-UHFFFAOYSA-N | $4.4\times10^5$ | | Wang et al. (2017) | Q | 80, 240 |
| MCM:HO3C96OOH | $8.0\times10^3$ | | Wang et al. (2017) | Q | 80, 238 |
| $C_9H_{20}O_3$ | $1.9\times10^4$ | | Wang et al. (2017) | Q | 80, 239 |
| GGTTUTGCXALAOI-UHFFFAOYSA-N | $8.9\times10^2$ | | Wang et al. (2017) | Q | 80, 240 |
| MCM:HO7C8CO3H | $1.0\times10^5$ | | Wang et al. (2017) | Q | 80, 238 |
| $C_9H_{18}O_4$ | $2.6\times10^4$ | | Wang et al. (2017) | Q | 80, 239 |
| LZPDGAGYXQEGIA-UHFFFAOYSA-N | 9.6 | | Wang et al. (2017) | Q | 80, 240 |
| MCM:HO8C9OOH | $8.9\times10^3$ | | Wang et al. (2017) | Q | 80, 238 |
| $C_9H_{20}O_3$ | $3.8\times10^2$ | | Wang et al. (2017) | Q | 80, 239 |
| DDKPTZKAQLYQQK-UHFFFAOYSA-N | $6.6\times10^2$ | | Wang et al. (2017) | Q | 80, 240 |
| MCM:IPBENZOLOH | $2.6\times10^{10}$ | | Wang et al. (2017) | Q | 80, 238 |
| $C_9H_{14}O_5$ | $1.0\times10^8$ | | Wang et al. (2017) | Q | 80, 239 |
| MXOPSFWTSRHQGT-UHFFFAOYSA-N | $1.5\times10^7$ | | Wang et al. (2017) | Q | 80, 240 |
| MCM:IPBZBPR2OH | $1.3\times10^7$ | | Wang et al. (2017) | Q | 80, 238 |
| $C_9H_{14}O_4$ | $6.8\times10^5$ | | Wang et al. (2017) | Q | 80, 239 |
| ZYFWWFFTFWUTQU-UHFFFAOYSA-N | $8.9\times10^5$ | | Wang et al. (2017) | Q | 80, 240 |





Table A3.4: Peroxides (ROOH) and peroxy radicals (ROO) (... continued)

| Substance Formula (Trivial Name) [CAS Registry Number] InChIKey | $H_s^{cp}$ (at $T^{\ominus}$) $\left[\dfrac{\mathrm{mol}}{\mathrm{m^3\,Pa}}\right]$ | $\dfrac{\mathrm{d}\ln H_s^{cp}}{\mathrm{d}(1/T)}$ [K] | Reference | Type | Note |
|---|---|---|---|---|---|
| MCM:IPBZBPROOH | $2.8\times10^8$ | | Wang et al. (2017) | Q | 80, 238 |
| $C_9H_{14}O_5$ | $2.6\times10^6$ | | Wang et al. (2017) | Q | 80, 239 |
| PFZQQVVUSVRBJH-UHFFFAOYSA-N | $2.5\times10^5$ | | Wang et al. (2017) | Q | 80, 240 |
| MCM:IPBZOLOOH | $5.9\times10^{11}$ | | Wang et al. (2017) | Q | 80, 238 |
| $C_9H_{14}O_6$ | $4.9\times10^7$ | | Wang et al. (2017) | Q | 80, 239 |
| CHCZSTHNGRHXBM-UHFFFAOYSA-N | $3.6\times10^6$ | | Wang et al. (2017) | Q | 80, 240 |
| MCM:METLBPR2OH | $7.4\times10^6$ | | Wang et al. (2017) | Q | 80, 238 |
| $C_9H_{14}O_4$ | $2.8\times10^5$ | | Wang et al. (2017) | Q | 80, 239 |
| WBMKIPDFTCRXQQ-UHFFFAOYSA-N | $5.3\times10^6$ | | Wang et al. (2017) | Q | 80, 240 |
| MCM:METLBPROOH | $1.7\times10^8$ | | Wang et al. (2017) | Q | 80, 238 |
| $C_9H_{14}O_5$ | $1.0\times10^6$ | | Wang et al. (2017) | Q | 80, 239 |
| JDOROKVPYHJTEM-UHFFFAOYSA-N | $4.9\times10^5$ | | Wang et al. (2017) | Q | 80, 240 |
| MCM:METOLOH | $1.3\times10^{10}$ | | Wang et al. (2017) | Q | 80, 238 |
| $C_9H_{14}O_5$ | $5.0\times10^7$ | | Wang et al. (2017) | Q | 80, 239 |
| JGSZPUTWQXTPGZ-UHFFFAOYSA-N | $5.1\times10^6$ | | Wang et al. (2017) | Q | 80, 240 |
| MCM:METOLOOH | $3.5\times10^{11}$ | | Wang et al. (2017) | Q | 80, 238 |
| $C_9H_{14}O_6$ | $2.3\times10^7$ | | Wang et al. (2017) | Q | 80, 239 |
| WHWNZVSDBMWGPT-UHFFFAOYSA-N | $2.0\times10^6$ | | Wang et al. (2017) | Q | 80, 240 |
| MCM:OETLBPR2OH | $7.4\times10^6$ | | Wang et al. (2017) | Q | 80, 238 |
| $C_9H_{14}O_4$ | $3.6\times10^5$ | | Wang et al. (2017) | Q | 80, 239 |
| HUTNYAIOWRBWAA-UHFFFAOYSA-N | $7.3\times10^5$ | | Wang et al. (2017) | Q | 80, 240 |
| MCM:OETLBPROOH | $1.7\times10^8$ | | Wang et al. (2017) | Q | 80, 238 |
| $C_9H_{14}O_5$ | $1.1\times10^6$ | | Wang et al. (2017) | Q | 80, 239 |
| JWHYGBDQJAPXFD-UHFFFAOYSA-N | $1.4\times10^5$ | | Wang et al. (2017) | Q | 80, 240 |
| MCM:OETOLOH | $1.7\times10^{10}$ | | Wang et al. (2017) | Q | 80, 238 |
| $C_9H_{14}O_5$ | $7.6\times10^7$ | | Wang et al. (2017) | Q | 80, 239 |
| RKSMHAQFVDQZCV-UHFFFAOYSA-N | $5.1\times10^8$ | | Wang et al. (2017) | Q | 80, 240 |
| MCM:OETOLOOH | $4.4\times10^{11}$ | | Wang et al. (2017) | Q | 80, 238 |
| $C_9H_{14}O_6$ | $3.1\times10^7$ | | Wang et al. (2017) | Q | 80, 239 |
| HMIPOFAAVJMIHR-UHFFFAOYSA-N | $4.5\times10^6$ | | Wang et al. (2017) | Q | 80, 240 |
| MCM:PBENZOLOH | $2.1\times10^{10}$ | | Wang et al. (2017) | Q | 80, 238 |
| $C_9H_{14}O_5$ | $7.8\times10^7$ | | Wang et al. (2017) | Q | 80, 239 |
| PPOCUJCGHLIXAB-UHFFFAOYSA-N | $2.8\times10^7$ | | Wang et al. (2017) | Q | 80, 240 |
| MCM:PBENZOLOOH | $5.6\times10^{11}$ | | Wang et al. (2017) | Q | 80, 238 |
| $C_9H_{14}O_6$ | $3.7\times10^7$ | | Wang et al. (2017) | Q | 80, 239 |
| YYEAFIOCCPRRFC-UHFFFAOYSA-N | $9.8\times10^5$ | | Wang et al. (2017) | Q | 80, 240 |
| MCM:PBZBPER2OH | $1.1\times10^7$ | | Wang et al. (2017) | Q | 80, 238 |
| $C_9H_{14}O_4$ | $5.0\times10^5$ | | Wang et al. (2017) | Q | 80, 239 |
| NXHGWQBANLWNAY-UHFFFAOYSA-N | $1.7\times10^6$ | | Wang et al. (2017) | Q | 80, 240 |



Table A3.4: Peroxides (ROOH) and peroxy radicals (ROO) (. . . continued)

| Substance Formula (Trivial Name) [CAS Registry Number] InChIKey | $H_s^{cp}$ (at $T^\ominus$) $\left[\dfrac{\text{mol}}{\text{m}^3\,\text{Pa}}\right]$ | $\dfrac{\text{d}\ln H_s^{cp}}{\text{d}(1/T)}$ [K] | Reference | Type | Note |
|---|---|---|---|---|---|
| MCM:PBZBPEROOH | $2.8\times10^8$ | | Wang et al. (2017) | Q | 80, 238 |
| $C_9H_{14}O_5$ | $2.0\times10^6$ | | Wang et al. (2017) | Q | 80, 239 |
| BDTBCPZZIFVKES-UHFFFAOYSA-N | $2.0\times10^6$ | | Wang et al. (2017) | Q | 80, 240 |
| MCM:PETLBPR2OH | $7.4\times10^6$ | | Wang et al. (2017) | Q | 80, 238 |
| $C_9H_{14}O_4$ | $3.6\times10^5$ | | Wang et al. (2017) | Q | 80, 239 |
| ZJSMOJBPUXMKGO-UHFFFAOYSA-N | $2.8\times10^5$ | | Wang et al. (2017) | Q | 80, 240 |
| MCM:PETLBPROOH | $1.7\times10^8$ | | Wang et al. (2017) | Q | 80, 238 |
| $C_9H_{14}O_5$ | $1.1\times10^6$ | | Wang et al. (2017) | Q | 80, 239 |
| HYHORQQVDRWXSH-UHFFFAOYSA-N | $8.1\times10^5$ | | Wang et al. (2017) | Q | 80, 240 |
| MCM:PETOLOH | $1.7\times10^{10}$ | | Wang et al. (2017) | Q | 80, 238 |
| $C_9H_{14}O_5$ | $8.5\times10^7$ | | Wang et al. (2017) | Q | 80, 239 |
| WWMQWQMHOYSCPT-UHFFFAOYSA-N | $1.6\times10^6$ | | Wang et al. (2017) | Q | 80, 240 |
| MCM:PETOLOOH | $4.4\times10^{11}$ | | Wang et al. (2017) | Q | 80, 238 |
| $C_9H_{14}O_6$ | $3.6\times10^7$ | | Wang et al. (2017) | Q | 80, 239 |
| NWKQTACFFZGFMV-UHFFFAOYSA-N | $6.6\times10^5$ | | Wang et al. (2017) | Q | 80, 240 |
| MCM:TM123BP2OH | $4.6\times10^6$ | | Wang et al. (2017) | Q | 80, 238 |
| $C_9H_{14}O_4$ | $2.3\times10^5$ | | Wang et al. (2017) | Q | 80, 239 |
| XELNTWYHIBVYRB-UHFFFAOYSA-N | $1.3\times10^5$ | | Wang et al. (2017) | Q | 80, 240 |
| MCM:TM123BPOOH | $1.2\times10^8$ | | Wang et al. (2017) | Q | 80, 238 |
| $C_9H_{14}O_5$ | $8.7\times10^5$ | | Wang et al. (2017) | Q | 80, 239 |
| VYRKWTIFXGFOKI-UHFFFAOYSA-N | $1.6\times10^5$ | | Wang et al. (2017) | Q | 80, 240 |
| MCM:TM123OLOH | $1.4\times10^{10}$ | | Wang et al. (2017) | Q | 80, 238 |
| $C_9H_{14}O_5$ | $8.5\times10^7$ | | Wang et al. (2017) | Q | 80, 239 |
| CUPCKHHMKYENPF-UHFFFAOYSA-N | $7.8\times10^6$ | | Wang et al. (2017) | Q | 80, 240 |
| MCM:TM123OLOOH | $3.6\times10^{11}$ | | Wang et al. (2017) | Q | 80, 238 |
| $C_9H_{14}O_6$ | $3.8\times10^7$ | | Wang et al. (2017) | Q | 80, 239 |
| QKUAZMPWBBDUFL-UHFFFAOYSA-N | $3.3\times10^4$ | | Wang et al. (2017) | Q | 80, 240 |
| MCM:TM124BP2OH | $4.6\times10^6$ | | Wang et al. (2017) | Q | 80, 238 |
| $C_9H_{14}O_4$ | $2.3\times10^5$ | | Wang et al. (2017) | Q | 80, 239 |
| ALTMQDHYGSUKTA-UHFFFAOYSA-N | $1.5\times10^5$ | | Wang et al. (2017) | Q | 80, 240 |
| MCM:TM124BPOOH | $1.2\times10^8$ | | Wang et al. (2017) | Q | 80, 238 |
| $C_9H_{14}O_5$ | $7.1\times10^5$ | | Wang et al. (2017) | Q | 80, 239 |
| WAGRVRZIKOHUQN-UHFFFAOYSA-N | $2.3\times10^5$ | | Wang et al. (2017) | Q | 80, 240 |
| MCM:TM124OLOH | $1.4\times10^{10}$ | | Wang et al. (2017) | Q | 80, 238 |
| $C_9H_{14}O_5$ | $1.2\times10^8$ | | Wang et al. (2017) | Q | 80, 239 |
| BDZMWXRIFUJLAS-UHFFFAOYSA-N | $1.0\times10^6$ | | Wang et al. (2017) | Q | 80, 240 |
| MCM:TM124OLOOH | $3.6\times10^{11}$ | | Wang et al. (2017) | Q | 80, 238 |
| $C_9H_{14}O_6$ | $5.4\times10^7$ | | Wang et al. (2017) | Q | 80, 239 |
| BWEFFYCDYKVJMY-UHFFFAOYSA-N | $2.0\times10^5$ | | Wang et al. (2017) | Q | 80, 240 |





Table A3.4: Peroxides (ROOH) and peroxy radicals (ROO) (. . . continued)

| Substance<br>Formula<br>(Trivial Name)<br>[CAS Registry Number]<br>InChIKey | $H_s^{cp}$<br>(at $T^{\ominus}$)<br>$\left[\dfrac{\mathrm{mol}}{\mathrm{m^3\,Pa}}\right]$ | $\dfrac{\mathrm{d}\ln H_s^{cp}}{\mathrm{d}(1/T)}$<br><br>[K] | Reference | Type | Note |
|---|---|---|---|---|---|
| MCM:TM135BP2OH | $5.6\times10^6$ | | Wang et al. (2017) | Q | 80, 238 |
| $C_9H_{14}O_4$ | $2.8\times10^5$ | | Wang et al. (2017) | Q | 80, 239 |
| UOFFRVMQLZOQAS-UHFFFAOYSA-N | $1.2\times10^6$ | | Wang et al. (2017) | Q | 80, 240 |
| MCM:TM135BPOOH | $1.4\times10^8$ | | Wang et al. (2017) | Q | 80, 238 |
| $C_9H_{14}O_5$ | $7.3\times10^5$ | | Wang et al. (2017) | Q | 80, 239 |
| JMJJTOHEWPZJQK-UHFFFAOYSA-N | $1.0\times10^6$ | | Wang et al. (2017) | Q | 80, 240 |
| MCM:TM135OLOH | $1.1\times10^{10}$ | | Wang et al. (2017) | Q | 80, 238 |
| $C_9H_{14}O_5$ | $5.4\times10^7$ | | Wang et al. (2017) | Q | 80, 239 |
| QLKJHZCZIQIKHZ-UHFFFAOYSA-N | $2.1\times10^6$ | | Wang et al. (2017) | Q | 80, 240 |
| MCM:TM135OLOOH | $2.9\times10^{11}$ | | Wang et al. (2017) | Q | 80, 238 |
| $C_9H_{14}O_6$ | $2.1\times10^7$ | | Wang et al. (2017) | Q | 80, 239 |
| BLJZZZVPECPSDH-UHFFFAOYSA-N | $6.5\times10^5$ | | Wang et al. (2017) | Q | 80, 240 |
| MCM:APINAOOH | $4.7\times10^4$ | | Wang et al. (2017) | Q | 80, 238 |
| $C_{10}H_{18}O_3$ | $1.5\times10^4$ | | Wang et al. (2017) | Q | 80, 239 |
| RYIWEMFTAFVTLU-UHFFFAOYSA-N | $1.3\times10^4$ | | Wang et al. (2017) | Q | 80, 240 |
| MCM:APINBOOH | $4.7\times10^4$ | | Wang et al. (2017) | Q | 80, 238 |
| $C_{10}H_{18}O_3$ | $1.8\times10^4$ | | Wang et al. (2017) | Q | 80, 239 |
| USRGRUOHOYDFID-UHFFFAOYSA-N | $1.3\times10^4$ | | Wang et al. (2017) | Q | 80, 240 |
| MCM:APINCOOH | $4.9\times10^4$ | | Wang et al. (2017) | Q | 80, 238 |
| $C_{10}H_{18}O_3$ | $1.6\times10^5$ | | Wang et al. (2017) | Q | 80, 239 |
| PONCTWLYBMTTOF-UHFFFAOYSA-N | $3.6\times10^4$ | | Wang et al. (2017) | Q | 80, 240 |
| MCM:BPINAOOH | $4.1\times10^4$ | | Wang et al. (2017) | Q | 80, 238 |
| $C_{10}H_{18}O_3$ | $1.6\times10^3$ | | Wang et al. (2017) | Q | 80, 239 |
| YIWAKBLZYMRLPO-UHFFFAOYSA-N | $4.9\times10^3$ | | Wang et al. (2017) | Q | 80, 240 |
| MCM:BPINBOOH | $4.1\times10^4$ | | Wang et al. (2017) | Q | 80, 238 |
| $C_{10}H_{18}O_3$ | $1.2\times10^3$ | | Wang et al. (2017) | Q | 80, 239 |
| LNOABRBKZGHYLB-UHFFFAOYSA-N | $5.8\times10^3$ | | Wang et al. (2017) | Q | 80, 240 |
| MCM:BPINCOOH | $4.6\times10^4$ | | Wang et al. (2017) | Q | 80, 238 |
| $C_{10}H_{18}O_3$ | $2.7\times10^5$ | | Wang et al. (2017) | Q | 80, 239 |
| ONJNLKNHLLSXIR-UHFFFAOYSA-N | $3.0\times10^5$ | | Wang et al. (2017) | Q | 80, 240 |
| MCM:C918CO3H | $2.2\times10^4$ | | Wang et al. (2017) | Q | 80, 238 |
| $C_{10}H_{16}O_4$ | $5.9\times10^1$ | | Wang et al. (2017) | Q | 80, 239 |
| NCSYYALRPIRWRM-UHFFFAOYSA-N | $4.9\times10^1$ | | Wang et al. (2017) | Q | 80, 240 |
| MCM:DMEBPR2OH | $5.0\times10^6$ | | Wang et al. (2017) | Q | 80, 238 |
| $C_{10}H_{16}O_4$ | $2.0\times10^5$ | | Wang et al. (2017) | Q | 80, 239 |
| JNPNKLHEXQFLDW-UHFFFAOYSA-N | $4.4\times10^5$ | | Wang et al. (2017) | Q | 80, 240 |
| MCM:DMEBPROOH | $1.2\times10^8$ | | Wang et al. (2017) | Q | 80, 238 |
| $C_{10}H_{16}O_5$ | $7.4\times10^5$ | | Wang et al. (2017) | Q | 80, 239 |
| FGEXZPMCUOXGAY-UHFFFAOYSA-N | $3.0\times10^5$ | | Wang et al. (2017) | Q | 80, 240 |





Table A3.4: Peroxides (ROOH) and peroxy radicals (ROO) (. . . continued)

| Substance Formula (Trivial Name) [CAS Registry Number] InChIKey | $H_s^{cp}$ (at $T^{\ominus}$) $\left[\dfrac{\text{mol}}{\text{m}^3\,\text{Pa}}\right]$ | $\dfrac{\mathrm{d}\ln H_s^{cp}}{\mathrm{d}(1/T)}$ [K] | Reference | Type | Note |
|---|---|---|---|---|---|
| MCM:DMEPHOLOH C$_{10}$H$_{16}$O$_5$ KIALIUKUKYONFN-UHFFFAOYSA-N | $8.9\times10^9$ $3.7\times10^7$ $4.9\times10^5$ | | Wang et al. (2017) Wang et al. (2017) Wang et al. (2017) | Q Q Q | 80, 238 80, 239 80, 240 |
| MCM:DMEPHOLOOH C$_{10}$H$_{16}$O$_6$ NQWKUASPEMANIL-UHFFFAOYSA-N | $2.3\times10^{11}$ $1.4\times10^7$ $3.2\times10^5$ | | Wang et al. (2017) Wang et al. (2017) Wang et al. (2017) | Q Q Q | 80, 238 80, 239 80, 240 |
| MCM:HO3C106OOH C$_{10}$H$_{22}$O$_3$ FUGKCLXVXWNPCD-UHFFFAOYSA-N | $6.5\times10^3$ $1.6\times10^4$ $1.1\times10^3$ | | Wang et al. (2017) Wang et al. (2017) Wang et al. (2017) | Q Q Q | 80, 238 80, 239 80, 240 |
| MCM:HO8C9CO3H C$_{10}$H$_{20}$O$_4$ UIPJRQQFQSAZPI-UHFFFAOYSA-N | $8.1\times10^4$ $2.5\times10^4$ $7.3$ | | Wang et al. (2017) Wang et al. (2017) Wang et al. (2017) | Q Q Q | 80, 238 80, 239 80, 240 |
| MCM:LIMAOOH C$_{10}$H$_{18}$O$_3$ FPMDSGWWEURSNO-UHFFFAOYSA-N | $3.6\times10^4$ $4.4\times10^4$ $2.7\times10^4$ $7.1\times10^3$ | 13000 | Wieser et al. (2023) Wang et al. (2017) Wang et al. (2017) Wang et al. (2017) | Q Q Q Q | 437 80, 238 80, 239 80, 240 |
| MCM:LIMBOOH C$_{10}$H$_{18}$O$_3$ KUZZIHAARVNWSI-UHFFFAOYSA-N | $4.4\times10^4$ $2.9\times10^4$ $2.0\times10^4$ | | Wang et al. (2017) Wang et al. (2017) Wang et al. (2017) | Q Q Q | 80, 238 80, 239 80, 240 |
| MCM:LIMCOOH C$_{10}$H$_{18}$O$_3$ VVHMRXSQYYKNLC-UHFFFAOYSA-N | $3.6\times10^4$ $5.3\times10^4$ $1.0\times10^4$ $4.5\times10^3$ | 13000 | Wieser et al. (2023) Wang et al. (2017) Wang et al. (2017) Wang et al. (2017) | Q Q Q Q | 437 80, 238 80, 239 80, 240 |
| MCM:DEMPHOLOH C$_{11}$H$_{18}$O$_5$ ZTCJTCVANLKQAR-UHFFFAOYSA-N | $8.0\times10^9$ $2.3\times10^7$ $1.4\times10^5$ | | Wang et al. (2017) Wang et al. (2017) Wang et al. (2017) | Q Q Q | 80, 238 80, 239 80, 240 |
| MCM:DEMPHOLOOH C$_{11}$H$_{18}$O$_6$ MGZZNFCOXGVYJN-UHFFFAOYSA-N | $2.1\times10^{11}$ $1.1\times10^7$ $1.4\times10^5$ | | Wang et al. (2017) Wang et al. (2017) Wang et al. (2017) | Q Q Q | 80, 238 80, 239 80, 240 |
| MCM:DETLBPR2OH C$_{11}$H$_{18}$O$_4$ DLRWBBGTDDBDMY-UHFFFAOYSA-N | $4.2\times10^6$ $1.7\times10^5$ $8.5\times10^5$ | | Wang et al. (2017) Wang et al. (2017) Wang et al. (2017) | Q Q Q | 80, 238 80, 239 80, 240 |
| MCM:DETLBPROOH C$_{11}$H$_{18}$O$_5$ XWTOJCWARUYVPA-UHFFFAOYSA-N | $1.0\times10^8$ $4.2\times10^5$ $1.1\times10^6$ | | Wang et al. (2017) Wang et al. (2017) Wang et al. (2017) | Q Q Q | 80, 238 80, 239 80, 240 |
| MCM:HO3C116OOH C$_{11}$H$_{24}$O$_3$ XCBVDNPGCYSIMS-UHFFFAOYSA-N | $5.8\times10^3$ $1.3\times10^4$ $1.5\times10^2$ | | Wang et al. (2017) Wang et al. (2017) Wang et al. (2017) | Q Q Q | 80, 238 80, 239 80, 240 |
| MCM:HO3C126OOH C$_{12}$H$_{26}$O$_3$ FENUAUYSPLOZDX-UHFFFAOYSA-N | $4.8\times10^3$ $1.0\times10^4$ $4.7\times10^2$ | | Wang et al. (2017) Wang et al. (2017) Wang et al. (2017) | Q Q Q | 80, 238 80, 239 80, 240 |



Table A3.4: Peroxides (ROOH) and peroxy radicals (ROO) (. . . continued)

| Substance<br>Formula<br>(Trivial Name)<br>[CAS Registry Number]<br>InChIKey | $H_s^{cp}$<br>(at $T^{\ominus}$)<br>$\left[\dfrac{\mathrm{mol}}{\mathrm{m^3\,Pa}}\right]$ | $\dfrac{\mathrm{d\ln} H_s^{cp}}{\mathrm{d}(1/T)}$<br><br>[K] | Reference | Type | Note |
|---|---|---|---|---|---|
| MCM:BCAOOH | $4.0\times10^4$ | | Wang et al. (2017) | Q | 80, 238 |
| $C_{15}H_{26}O_3$ | $3.6\times10^3$ | | Wang et al. (2017) | Q | 80, 239 |
| UHQMVBPMTRQRFN-UHFFFAOYSA-N | $1.0\times10^5$ | | Wang et al. (2017) | Q | 80, 240 |
| MCM:BCBOOH | $4.0\times10^4$ | | Wang et al. (2017) | Q | 80, 238 |
| $C_{15}H_{26}O_3$ | $1.0\times10^4$ | | Wang et al. (2017) | Q | 80, 239 |
| LZWWDZWHHIWFFS-UHFFFAOYSA-N | $6.3\times10^4$ | | Wang et al. (2017) | Q | 80, 240 |
| MCM:BCCOOH | $5.8\times10^4$ | | Wang et al. (2017) | Q | 80, 238 |
| $C_{15}H_{26}O_3$ | $3.7\times10^4$ | | Wang et al. (2017) | Q | 80, 239 |
| LYZHLRDUTRINQK-UHFFFAOYSA-N | $2.5\times10^4$ | | Wang et al. (2017) | Q | 80, 240 |
| MCM:CATEC1OOH | $7.4\times10^4$ | | Wang et al. (2017) | Q | 80, 238 |
| $C_6H_6O_3$ | $1.4\times10^3$ | | Wang et al. (2017) | Q | 80, 239 |
| GQLFYNLDVCYDKN-UHFFFAOYSA-N | $2.5\times10^2$ | | Wang et al. (2017) | Q | 80, 240 |
| MCM:MCATEC1OOH | $4.6\times10^4$ | | Wang et al. (2017) | Q | 80, 238 |
| $C_7H_8O_3$ | $1.6\times10^3$ | | Wang et al. (2017) | Q | 80, 239 |
| XVDDPRZRYAGIKB-UHFFFAOYSA-N | $2.2\times10^2$ | | Wang et al. (2017) | Q | 80, 240 |
| MCM:ECATEC1OOH | $4.0\times10^4$ | | Wang et al. (2017) | Q | 80, 238 |
| $C_8H_{10}O_3$ | $8.3\times10^2$ | | Wang et al. (2017) | Q | 80, 239 |
| WXEIAPGLSOOSMJ-UHFFFAOYSA-N | $5.3\times10^2$ | | Wang et al. (2017) | Q | 80, 240 |
| MCM:MXCTEC1OOH | $3.0\times10^4$ | | Wang et al. (2017) | Q | 80, 238 |
| $C_8H_{10}O_3$ | $1.6\times10^3$ | | Wang et al. (2017) | Q | 80, 239 |
| OTSRNDWSIFNBPX-UHFFFAOYSA-N | $8.3\times10^6$ | | Wang et al. (2017) | Q | 80, 240 |
| MCM:OCATEC1OOH | $3.0\times10^4$ | | Wang et al. (2017) | Q | 80, 238 |
| $C_8H_{10}O_3$ | $1.7\times10^3$ | | Wang et al. (2017) | Q | 80, 239 |
| FFFASTPMSYAWLR-UHFFFAOYSA-N | $1.4\times10^3$ | | Wang et al. (2017) | Q | 80, 240 |
| MCM:PCATEC1OOH | $3.0\times10^4$ | | Wang et al. (2017) | Q | 80, 238 |
| $C_8H_{10}O_3$ | $3.1\times10^3$ | | Wang et al. (2017) | Q | 80, 239 |
| XEPXMQBYFZNYBF-UHFFFAOYSA-N | $5.6\times10^1$ | | Wang et al. (2017) | Q | 80, 240 |
| MCM:STYRENOOH | $3.2\times10^6$ | | Wang et al. (2017) | Q | 80, 238 |
| $C_8H_{10}O_3$ | $2.1\times10^4$ | | Wang et al. (2017) | Q | 80, 239 |
| PRSNAZGJNCSLPW-UHFFFAOYSA-N | $4.4\times10^4$ | | Wang et al. (2017) | Q | 80, 240 |
| MCM:IPCATC1OOH | $3.8\times10^4$ | | Wang et al. (2017) | Q | 80, 238 |
| $C_9H_{12}O_3$ | $6.0\times10^2$ | | Wang et al. (2017) | Q | 80, 239 |
| LUGXJULPVOELPW-UHFFFAOYSA-N | $5.8\times10^2$ | | Wang et al. (2017) | Q | 80, 240 |
| MCM:MTCTEC1OOH | $2.3\times10^4$ | | Wang et al. (2017) | Q | 80, 238 |
| $C_9H_{12}O_3$ | $7.6\times10^2$ | | Wang et al. (2017) | Q | 80, 239 |
| RXVYKZMASOKYNM-UHFFFAOYSA-N | $6.5\times10^6$ | | Wang et al. (2017) | Q | 80, 240 |
| MCM:OTCTEC1OOH | $2.3\times10^4$ | | Wang et al. (2017) | Q | 80, 238 |
| $C_9H_{12}O_3$ | $8.9\times10^2$ | | Wang et al. (2017) | Q | 80, 239 |
| BNYRTZAXTAHDLR-UHFFFAOYSA-N | $8.9\times10^2$ | | Wang et al. (2017) | Q | 80, 240 |



Table A3.4: Peroxides (ROOH) and peroxy radicals (ROO) (. . . continued)

| Substance Formula (Trivial Name) [CAS Registry Number] InChIKey | $H_s^{cp}$ (at $T^\ominus$) $\left[\dfrac{\text{mol}}{\text{m}^3\,\text{Pa}}\right]$ | $\dfrac{\text{d}\ln H_s^{cp}}{\text{d}(1/T)}$ [K] | Reference | Type | Note |
|---|---|---|---|---|---|
| MCM:PRCATC1OOH | $3.2\times10^4$ | | Wang et al. (2017) | Q | 80, 238 |
| $C_9H_{12}O_3$ | $5.9\times10^2$ | | Wang et al. (2017) | Q | 80, 239 |
| WDHGVTVSFNORIO-UHFFFAOYSA-N | $6.0\times10^2$ | | Wang et al. (2017) | Q | 80, 240 |
| MCM:PTCTEC1OOH | $2.3\times10^4$ | | Wang et al. (2017) | Q | 80, 238 |
| $C_9H_{12}O_3$ | $1.6\times10^3$ | | Wang et al. (2017) | Q | 80, 239 |
| SXYSFXQBYZNKPP-UHFFFAOYSA-N | $4.5\times10^1$ | | Wang et al. (2017) | Q | 80, 240 |
| MCM:T123CT1OOH | $1.7\times10^4$ | | Wang et al. (2017) | Q | 80, 238 |
| $C_9H_{12}O_3$ | $2.0\times10^3$ | | Wang et al. (2017) | Q | 80, 239 |
| XFULZEJTRUOVEV-UHFFFAOYSA-N | $7.3\times10^6$ | | Wang et al. (2017) | Q | 80, 240 |
| MCM:T124CT1OOH | $1.7\times10^4$ | | Wang et al. (2017) | Q | 80, 238 |
| $C_9H_{12}O_3$ | $3.2\times10^3$ | | Wang et al. (2017) | Q | 80, 239 |
| XVSFIBYJDMNABQ-UHFFFAOYSA-N | $2.7\times10^6$ | | Wang et al. (2017) | Q | 80, 240 |
| MCM:HCOCH2OOH | $7.4\times10^3$ | | Wang et al. (2017) | Q | 80, 238 |
| $C_2H_4O_3$ | $1.4\times10^2$ | | Wang et al. (2017) | Q | 80, 239 |
| TUJPSEFVDYHSJJ-UHFFFAOYSA-N | $4.2\times10^1$ | | Wang et al. (2017) | Q | 80, 240 |
| MCM:HCOCO3H | $9.8\times10^4$ | | Wang et al. (2017) | Q | 80, 238 |
| $C_2H_2O_4$ | $3.9\times10^2$ | | Wang et al. (2017) | Q | 80, 239 |
| JOALXJIWVKUVBR-UHFFFAOYSA-N | $1.6\times10^1$ | | Wang et al. (2017) | Q | 80, 240 |
| MCM:C3DIALOOH | $5.6\times10^6$ | | Wang et al. (2017) | Q | 80, 238 |
| $C_3H_4O_4$ | $1.7\times10^4$ | | Wang et al. (2017) | Q | 80, 239 |
| VUPDPJIDXKCVGY-UHFFFAOYSA-N | $4.7\times10^1$ | | Wang et al. (2017) | Q | 80, 240 |
| MCM:CHOC2H4OOH | $5.8\times10^3$ | | Wang et al. (2017) | Q | 80, 238 |
| $C_3H_6O_3$ | $5.3\times10^3$ | | Wang et al. (2017) | Q | 80, 239 |
| XSASRUDTFFBDDK-UHFFFAOYSA-N | $1.6\times10^3$ | | Wang et al. (2017) | Q | 80, 240 |



### A3.5   Aldehydes (RCHO)

Table A3.5: Aldehydes (RCHO)

| Substance<br>Formula<br>(Trivial Name)<br>[CAS Registry Number]<br>InChIKey | $H_s^{cp}$<br>(at $T^{\ominus}$)<br>$\left[\dfrac{\text{mol}}{\text{m}^3\,\text{Pa}}\right]$ | $\dfrac{\text{d}\ln H_s^{cp}}{\text{d}(1/T)}$<br><br>[K] | Reference | Type | Note |
|---|---|---|---|---|---|
| methanal | $3.2\times10^1$ | 7100 | Burkholder et al. (2019) | L | 455 |
| HCHO | $3.2\times10^1$ | 7100 | Burkholder et al. (2015) | L | 455 |
| (formaldehyde) | $3.2\times10^1$ | 6800 | Warneck and Williams (2012) | L | 455 |
| [50-00-0] | $3.2\times10^1$ | 7100 | Sander et al. (2011) | L | 455 |
| WSFSSNUMVMOOMR-UHFFFAOYSA-N | $3.2\times10^1$ | 7100 | Sander et al. (2006) | L | 455 |
| | $3.2\times10^1$ | 6800 | Staudinger and Roberts (2001) | L | 455 |
| | $3.2\times10^1$ | 6800 | Staudinger and Roberts (1996) | L | 455 |
| | $3.5\times10^1$ | 5700 | Liu et al. (2015) | M | 455 |
| | $3.4\times10^1$ | 6400 | Allou et al. (2011) | M | 455 |
| | $5.3\times10^1$ | 1600 | Seyfioglu and Odabasi (2007) | M | 455 |
| | $9.9\times10^1$ | | Kim et al. (2000) | M | 87, 455 |
| | $3.1\times10^1$ | 6500 | Zhou and Mopper (1990) | M | 456, 455 |
| | $3.1\times10^1$ | 7200 | Betterton and Hoffmann (1988) | M | 455 |
| | | | Dong and Dasgupta (1986) | M | 457 |
| | | | Ledbury and Blair (1925) | M | 458 |
| | | | Blair and Ledbury (1925) | M | 458 |
| | $3.0\times10^1$ | | Lide and Frederikse (1995) | V | 455 |
| | 2.3 | | Hwang et al. (1992) | V | 455 |
| | $6.9\times10^1$ | 6400 | Chameides (1984) | T | 455 |
| | $2.9\times10^1$ | 7200 | Bell (1966) | X | 459, 455 |
| | $5.9\times10^1$ | | Gaffney and Senum (1984) | X | 455, 389 |
| | $4.5\times10^1$ | | Lee and Zhou (1993) | C | 87, 455 |
| | | | Hough (1991) | C | 458 |
| | $1.4\times10^2$ | | Warneck (1988) | C | 455 |
| | $7.8\times10^{-2}$ | | Wang et al. (2017) | Q | 80, 238 |
| | $4.9\times10^{-2}$ | | Wang et al. (2017) | Q | 80, 239 |
| | $1.0\times10^{-2}$ | | Wang et al. (2017) | Q | 80, 240 |
| | $2.8\times10^{-2}$ | | Hilal et al. (2008) | Q | |
| | $7.5\times10^{-2}$ | | Modarresi et al. (2007) | Q | 67 |
| | $4.3\times10^{-2}$ | | Yaffe et al. (2003) | Q | 248, 249 |
| | $9.0\times10^{-2}$ | | English and Carroll (2001) | Q | 230, 231 |
| | $6.2\times10^{-2}$ | | Katritzky et al. (1998) | Q | |
| | $1.8\times10^{-1}$ | | Nirmalakhandan et al. (1997) | Q | |
| | $1.0\times10^2$ | | Meylan and Howard (1991) | Q | 455 |
| | $4.2\times10^{-2}$ | | Abraham et al. (1990) | ? | |
| | $6.2\times10^1$ | | Seinfeld (1986) | ? | 21, 455 |
| | | | Lelieveld and Crutzen (1991) | W | 458 |
| | | | Pandis and Seinfeld (1989) | W | 458 |



Table A3.5: Aldehydes (RCHO) (...continued)

| Substance Formula (Trivial Name) [CAS Registry Number] InChIKey | $H_s^{cp}$ (at $T^\ominus$) $\left[\dfrac{\mathrm{mol}}{\mathrm{m^3\,Pa}}\right]$ | $\dfrac{\mathrm{d}\ln H_s^{cp}}{\mathrm{d}(1/T)}$ [K] | Reference | Type | Note |
|---|---|---|---|---|---|
| ethanal | $1.3\times10^{-1}$ | 5900 | Burkholder et al. (2019) | L | 460 |
| $CH_3CHO$ | $1.3\times10^{-1}$ | 5900 | Burkholder et al. (2015) | L | 460 |
| (acetaldehyde) | $1.5\times10^{-1}$ | 5600 | Brockbank (2013) | L | 1 |
| [75-07-0] | $1.3\times10^{-1}$ | 5900 | Sander et al. (2011) | L | |
| IKHGUXGNUITLKF-UHFFFAOYSA-N | $1.3\times10^{-1}$ | 5900 | Sander et al. (2006) | L | |
| | $1.3\times10^{-1}$ | 5700 | Staudinger and Roberts (2001) | L | |
| | $1.4\times10^{-1}$ | 5600 | Staudinger and Roberts (1996) | L | |
| | $1.7\times10^{-1}$ | 5600 | Wieland et al. (2015) | M | 461 |
| | $1.5\times10^{-1}$ | 6400 | Ji and Evans (2007) | M | |
| | $1.1\times10^{-1}$ | | Straver and de Loos (2005) | M | |
| | $1.5\times10^{-1}$ | | Marin et al. (1999) | M | |
| | $1.3\times10^{-1}$ | 5700 | Benkelberg et al. (1995) | M | |
| | $1.7\times10^{-1}$ | 5000 | Zhou and Mopper (1990) | M | 456 |
| | $7.1\times10^{-2}$ | | Guitart et al. (1989) | M | 14 |
| | $1.2\times10^{-1}$ | 6300 | Betterton and Hoffmann (1988) | M | 460 |
| | $1.2\times10^{-1}$ | 5800 | Snider and Dawson (1985) | M | |
| | $8.3\times10^{-2}$ | | Richon et al. (1985) | M | 38 |
| | $1.6\times10^{-1}$ | | Mazza (1980) | M | |
| | $2.5\times10^{-1}$ | | Vitenberg et al. (1974) | M | 373 |
| | $1.5\times10^{-1}$ | | Buttery et al. (1969) | M | |
| | $1.2\times10^{-1}$ | | Marin et al. (1999) | V | |
| | $1.2\times10^{-1}$ | | Hwang et al. (1992) | V | |
| | $7.8\times10^{-2}$ | | Yaws (2003) | X | 258 |
| | $1.7\times10^{-2}$ | 4500 | Janini and Quaddora (1986) | X | 298 |
| | $1.7\times10^{-1}$ | 4700 | Goldstein (1982) | X | 298 |
| | $1.5\times10^{-1}$ | | Gaffney and Senum (1984) | X | 389 |
| | $1.5\times10^{-1}$ | | Pierotti et al. (1959) | X | 462 |
| | $1.8\times10^{-1}$ | | Dupeux et al. (2022) | Q | 259 |
| | $9.0\times10^{-2}$ | | Keshavarz et al. (2022) | Q | |
| | $9.8\times10^{-2}$ | | Duchowicz et al. (2020) | Q | |
| | $9.8\times10^{-2}$ | | Wang et al. (2017) | Q | 80, 238 |
| | $3.4\times10^{-1}$ | | Wang et al. (2017) | Q | 80, 239 |
| | $1.3\times10^{-1}$ | | Wang et al. (2017) | Q | 80, 240 |
| | $1.5\times10^{-1}$ | | Li et al. (2014) | Q | 241 |
| | $1.6\times10^{-1}$ | | Raventos-Duran et al. (2010) | Q | 242, 243 |
| | $3.9\times10^{-1}$ | | Raventos-Duran et al. (2010) | Q | 244 |
| | $1.6\times10^{-1}$ | | Raventos-Duran et al. (2010) | Q | 245 |
| | $1.1\times10^{-1}$ | | Hilal et al. (2008) | Q | |
| | $1.0\times10^{-1}$ | | Modarresi et al. (2007) | Q | 67 |
| | | 5200 | Kühne et al. (2005) | Q | |
| | $1.5\times10^{-1}$ | | Yaffe et al. (2003) | Q | 248, 249 |
| | $6.6\times10^{-2}$ | | Yao et al. (2002) | Q | 229 |
| | $1.4\times10^{-1}$ | | English and Carroll (2001) | Q | 230, 231 |
| | $1.4\times10^{-1}$ | | Marin et al. (1999) | Q | |
| | $7.7\times10^{-2}$ | | Katritzky et al. (1998) | Q | |
| | $1.5\times10^{-1}$ | | Nirmalakhandan et al. (1997) | Q | |
| | $1.3\times10^{-1}$ | | Suzuki et al. (1992) | Q | 232 |





Table A3.5: Aldehydes (RCHO) (...continued)

| Substance Formula (Trivial Name) [CAS Registry Number] InChIKey | $H_s^{cp}$ (at $T^\ominus$) $\left[\dfrac{\text{mol}}{\text{m}^3\,\text{Pa}}\right]$ | $\dfrac{\text{d}\ln H_s^{cp}}{\text{d}(1/T)}$ [K] | Reference | Type | Note |
|---|---|---|---|---|---|
| | $1.5\times10^{-1}$ | | Duchowicz et al. (2020) | ? | 185, 21 |
| | $1.5\times10^{-1}$ | | Mackay et al. (2006c) | ? | 21 |
| | | 5800 | Kühne et al. (2005) | ? | |
| | $1.0\times10^{-1}$ | | Yaws (1999) | ? | 21 |
| | $9.8\times10^{-2}$ | | Yaws and Yang (1992) | ? | 21 |
| | $1.5\times10^{-1}$ | | Abraham et al. (1990) | ? | |
| ethanedial | $4.1\times10^{3}$ | 7500 | Burkholder et al. (2019) | L | 460 |
| OHCCHO | $4.1\times10^{3}$ | 7500 | Burkholder et al. (2015) | L | 460 |
| (glyoxal) | $4.1\times10^{3}$ | 7500 | Sander et al. (2011) | L | 460 |
| [107-22-2] | $4.9\times10^{5}$ | | Kampf et al. (2013) | M | 460, 463 |
| LEQAOMBKQFMDFZ-UHFFFAOYSA-N | $4.1\times10^{3}$ | 7500 | Ip et al. (2009) | M | 460 |
| | | | Volkamer et al. (2009) | M | 464 |
| | $2.6\times10^{5}$ | | Kroll et al. (2005) | M | 460, 465 |
| | $3.6\times10^{3}$ | | Zhou and Mopper (1990) | M | 460, 70 |
| | $>3.0\times10^{3}$ | | Betterton and Hoffmann (1988) | M | 460 |
| | $1.4\times10^{4}$ | | Lee and Zhou (1993) | C | 87, 460 |
| | $2.0\times10^{2}$ | | Keshavarz et al. (2022) | Q | |
| | $1.6\times10^{1}$ | | Duchowicz et al. (2020) | Q | 299 |
| | $8.9\times10^{1}$ | | Wang et al. (2017) | Q | 80, 238 |
| | 9.6 | | Wang et al. (2017) | Q | 80, 239 |
| | $1.2\times10^{-2}$ | | Wang et al. (2017) | Q | 80, 240 |
| | $3.1\times10^{4}$ | | Raventos-Duran et al. (2010) | Q | 242, 243 |
| | $2.0\times10^{3}$ | | Raventos-Duran et al. (2010) | Q | 244 |
| | $2.5\times10^{1}$ | | Raventos-Duran et al. (2010) | Q | 245 |
| | $3.0\times10^{3}$ | | Duchowicz et al. (2020) | ? | 185, 21 |
| | $2.7\times10^{5}$ | | Woo and McNeill (2015) | ? | 466 |
| propanal | $9.9\times10^{-2}$ | 4300 | Burkholder et al. (2019) | L | |
| $C_2H_5CHO$ | $9.9\times10^{-2}$ | 4300 | Burkholder et al. (2015) | L | |
| (propionaldehyde) | $1.3\times10^{-1}$ | 5500 | Brockbank (2013) | L | 1 |
| [123-38-6] | $9.9\times10^{-2}$ | 4300 | Sander et al. (2011) | L | |
| NBBJYMSMWIIQGU-UHFFFAOYSA-N | $9.9\times10^{-2}$ | 4300 | Sander et al. (2006) | L | |
| | $1.3\times10^{-1}$ | | Liu et al. (2015) | M | 72 |
| | $9.1\times10^{-2}$ | | Kim and Kim (2014) | M | |
| | $1.3\times10^{-1}$ | 5800 | Ji and Evans (2007) | M | |
| | $1.3\times10^{-1}$ | 5700 | Zhou and Mopper (1990) | M | 456 |
| | $6.1\times10^{-2}$ | | Richon et al. (1985) | M | 38 |
| | $1.2\times10^{-1}$ | | Mazza (1980) | M | |
| | $1.3\times10^{-1}$ | | Buttery et al. (1969) | M | |
| | $7.5\times10^{-2}$ | | Buttery et al. (1965) | M | |
| | $1.3\times10^{-1}$ | | Mackay et al. (2006c) | V | |
| | $1.3\times10^{-2}$ | | Mackay et al. (1995) | V | |
| | $3.2\times10^{-2}$ | 3200 | Djerki and Laub (1988) | V | |
| | $1.6\times10^{-1}$ | | Amoore and Buttery (1978) | V | |
| | $4.3\times10^{-2}$ | | Yaws (2003) | X | 258 |
| | $5.2\times10^{-2}$ | 5600 | Schaffer and Daubert (1969) | X | 298 |
| | $2.7\times10^{-2}$ | 2400 | Janini and Quaddora (1986) | X | 298 |



Table A3.5: Aldehydes (RCHO) (. . . continued)

| Substance Formula (Trivial Name) [CAS Registry Number] InChIKey | $H_s^{cp}$ (at $T^{\ominus}$) $\left[\dfrac{\text{mol}}{\text{m}^3\,\text{Pa}}\right]$ | $\dfrac{\text{d}\ln H_s^{cp}}{\text{d}(1/T)}$ [K] | Reference | Type | Note |
|---|---|---|---|---|---|
| | $9.8\times10^{-2}$ | | Dupeux et al. (2022) | Q | 259 |
| | $1.2\times10^{-1}$ | | Keshavarz et al. (2022) | Q | |
| | $9.9\times10^{-2}$ | | Duchowicz et al. (2020) | Q | 184 |
| | $8.0\times10^{-2}$ | | Wang et al. (2017) | Q | 80, 238 |
| | $2.4\times10^{-1}$ | | Wang et al. (2017) | Q | 80, 239 |
| | $8.0\times10^{-2}$ | | Wang et al. (2017) | Q | 80, 240 |
| | $1.2\times10^{-2}$ | | Gharagheizi et al. (2012) | Q | |
| | $1.2\times10^{-1}$ | | Raventos-Duran et al. (2010) | Q | 242, 243 |
| | $2.5\times10^{-1}$ | | Raventos-Duran et al. (2010) | Q | 244 |
| | $9.9\times10^{-2}$ | | Raventos-Duran et al. (2010) | Q | 245 |
| | $1.2\times10^{-1}$ | | Hilal et al. (2008) | Q | |
| | $1.4\times10^{-1}$ | | Modarresi et al. (2007) | Q | 67 |
| | | 5500 | Kühne et al. (2005) | Q | |
| | $1.4\times10^{-1}$ | | Yaffe et al. (2003) | Q | 248, 249 |
| | $1.1\times10^{-1}$ | | English and Carroll (2001) | Q | 230, 231 |
| | $5.8\times10^{-2}$ | | Katritzky et al. (1998) | Q | |
| | $1.2\times10^{-1}$ | | Nirmalakhandan et al. (1997) | Q | |
| | $7.3\times10^{-2}$ | | Russell et al. (1992) | Q | 279 |
| | $1.0\times10^{-1}$ | | Suzuki et al. (1992) | Q | 232 |
| | $1.3\times10^{-1}$ | | Duchowicz et al. (2020) | ? | 185, 21 |
| | $1.3\times10^{-1}$ | | Mackay et al. (2006c) | ? | 21 |
| | | 5000 | Kühne et al. (2005) | ? | |
| | $2.3\times10^{-1}$ | | Yaws (1999) | ? | 21 |
| | $1.3\times10^{-1}$ | | Abraham et al. (1990) | ? | |
| propanedial C$_3$H$_4$O$_2$ (malonaldehyde) [542-78-9] WSMYVTOQOOLQHP-UHFFFAOYSA-N | $7.3\times10^{1}$ $6.0\times10^{1}$ 4.1 | | Wang et al. (2017) Wang et al. (2017) Wang et al. (2017) | Q Q Q | 80, 238 80, 239 80, 240 |
| MCM:C32OH13CO C$_3$H$_4$O$_3$ NVXLIZQNSVLKPO-UHFFFAOYSA-N | $2.3\times10^{3}$ $5.5\times10^{3}$ 8.7 | | Wang et al. (2017) Wang et al. (2017) Wang et al. (2017) | Q Q Q | 80, 238 80, 239 80, 240 |
| MCM:CH3CHOHCHO C$_3$H$_6$O$_2$ BSABBBMNWQWLLU-UHFFFAOYSA-N | $1.1\times10^{1}$ $1.1\times10^{2}$ 3.1 | | Wang et al. (2017) Wang et al. (2017) Wang et al. (2017) | Q Q Q | 80, 238 80, 239 80, 240 |
| MCM:HOC2H4CHO C$_3$H$_6$O$_2$ AKXKFZDCRYJKTF-UHFFFAOYSA-N | $3.6\times10^{1}$ $2.2\times10^{2}$ $8.3\times10^{2}$ $6.6\times10^{1}$ | 9900 | Wieser et al. (2023) Wang et al. (2017) Wang et al. (2017) Wang et al. (2017) | Q Q Q Q | 437 80, 238 80, 239 80, 240 |





Table A3.5: Aldehydes (RCHO) (. . . continued)

| Substance Formula (Trivial Name) [CAS Registry Number] InChIKey | $H_s^{cp}$ (at $T^{\ominus}$) $\left[\dfrac{\mathrm{mol}}{\mathrm{m}^3\,\mathrm{Pa}}\right]$ | $\dfrac{\mathrm{d}\ln H_s^{cp}}{\mathrm{d}(1/T)}$ [K] | Reference | Type | Note |
|---|---|---|---|---|---|
| butanal | $9.5\times10^{-2}$ | 6200 | Burkholder et al. (2019) | L | |
| $C_3H_7CHO$ | $9.5\times10^{-2}$ | 6200 | Burkholder et al. (2015) | L | |
| (butyraldehyde) | $9.1\times10^{-2}$ | 6000 | Brockbank (2013) | L | 1 |
| [123-72-8] | $9.5\times10^{-2}$ | 6200 | Sander et al. (2011) | L | |
| ZTQSAGDEMFDKMZ-UHFFFAOYSA-N | $9.5\times10^{-2}$ | 6200 | Sander et al. (2006) | L | |
| | $6.1\times10^{-2}$ | | Kim and Kim (2014) | M | |
| | $8.9\times10^{-2}$ | 6200 | Ji and Evans (2007) | M | |
| | $9.5\times10^{-2}$ | 6200 | Zhou and Mopper (1990) | M | 456 |
| | $8.6\times10^{-2}$ | | Buttery et al. (1969) | M | |
| | $6.4\times10^{-2}$ | | Buttery et al. (1965) | M | |
| | $6.5\times10^{-2}$ | | Mackay et al. (2006c) | V | |
| | $6.5\times10^{-2}$ | | Mackay et al. (1995) | V | |
| | $1.0\times10^{-1}$ | | Hwang et al. (1992) | V | |
| | $8.7\times10^{-2}$ | 3500 | Djerki and Laub (1988) | V | |
| | $6.7\times10^{-2}$ | | Amoore and Buttery (1978) | V | |
| | $8.4\times10^{-2}$ | | Yaws (2003) | X | 258 |
| | $8.3\times10^{-2}$ | | Yaws (2003) | X | 237 |
| | $5.4\times10^{-2}$ | 4000 | Janini and Quaddora (1986) | X | 298 |
| | $6.8\times10^{-2}$ | | Dupeux et al. (2022) | Q | 259 |
| | $1.6\times10^{-1}$ | | Keshavarz et al. (2022) | Q | |
| | $9.9\times10^{-2}$ | | Duchowicz et al. (2020) | Q | |
| | $6.3\times10^{-2}$ | | Wang et al. (2017) | Q | 80, 238 |
| | $1.4\times10^{-1}$ | | Wang et al. (2017) | Q | 80, 239 |
| | $6.3\times10^{-2}$ | | Wang et al. (2017) | Q | 80, 240 |
| | $9.9\times10^{-2}$ | | Raventos-Duran et al. (2010) | Q | 242, 243 |
| | $1.2\times10^{-1}$ | | Raventos-Duran et al. (2010) | Q | 244 |
| | $7.8\times10^{-2}$ | | Raventos-Duran et al. (2010) | Q | 245 |
| | $5.8\times10^{-2}$ | | Gharagheizi et al. (2010) | Q | 246 |
| | $9.0\times10^{-2}$ | | Hilal et al. (2008) | Q | |
| | $1.1\times10^{-1}$ | | Modarresi et al. (2007) | Q | 67 |
| | | 5900 | Kühne et al. (2005) | Q | |
| | $6.7\times10^{-2}$ | | Yaffe et al. (2003) | Q | 248, 272 |
| | $5.3\times10^{-2}$ | | Yao et al. (2002) | Q | 229 |
| | $8.6\times10^{-2}$ | | English and Carroll (2001) | Q | 230, 231 |
| | $5.4\times10^{-2}$ | | Katritzky et al. (1998) | Q | |
| | $9.5\times10^{-2}$ | | Nirmalakhandan et al. (1997) | Q | |
| | $8.6\times10^{-2}$ | | Russell et al. (1992) | Q | 279 |
| | $7.9\times10^{-2}$ | | Suzuki et al. (1992) | Q | 232 |
| | $8.6\times10^{-2}$ | | Duchowicz et al. (2020) | ? | 185, 21 |
| | $8.6\times10^{-2}$ | | Mackay et al. (2006c) | ? | 21 |
| | | 6400 | Kühne et al. (2005) | ? | |
| | $8.4\times10^{-2}$ | | Yaws (1999) | ? | 21 |
| | $8.6\times10^{-2}$ | | Abraham et al. (1990) | ? | |



Table A3.5: Aldehydes (RCHO) (... continued)

| Substance / Formula / (Trivial Name) / [CAS Registry Number] / InChIKey | $H_s^{cp}$ (at $T^\ominus$) $\left[\dfrac{\text{mol}}{\text{m}^3\,\text{Pa}}\right]$ | $\dfrac{\text{d}\ln H_s^{cp}}{\text{d}(1/T)}$ [K] | Reference | Type | Note |
|---|---|---|---|---|---|
| 2-methylpropanal | $5.1\times10^{-2}$ | | Burkholder et al. (2019) | L | |
| $C_4H_8O$ | $5.1\times10^{-2}$ | | Brockbank (2013) | L | |
| (isobutyraldehyde) | $3.2\times10^{-2}$ | 7600 | Bruneel et al. (2016) | M | |
| [78-84-2] | $5.9\times10^{-3}$ | 4500 | Strekowski and George (2005) | M | |
| AMIMRNSIRUDHCM-UHFFFAOYSA-N | $3.3\times10^{-2}$ | | Karl et al. (2003) | M | |
| | $3.4\times10^{-2}$ | | Pollien et al. (2003) | M | |
| | $5.0\times10^{-2}$ | | Amoore and Buttery (1978) | M | |
| | $5.5\times10^{-2}$ | | Duchowicz et al. (2020) | V | 186 |
| | $5.5\times10^{-2}$ | | HSDB (2015) | V | |
| | $6.7\times10^{-2}$ | | Amoore and Buttery (1978) | V | |
| | $5.8\times10^{-2}$ | | Yaws (2003) | X | 258 |
| | $4.5\times10^{-2}$ | | Dupeux et al. (2022) | Q | 259 |
| | $3.8\times10^{-2}$ | | Duchowicz et al. (2020) | Q | |
| | $7.4\times10^{-2}$ | | Wang et al. (2017) | Q | 80, 238 |
| | $1.1\times10^{-1}$ | | Wang et al. (2017) | Q | 80, 239 |
| | $4.3\times10^{-2}$ | | Wang et al. (2017) | Q | 80, 240 |
| | $7.0\times10^{-2}$ | | Hilal et al. (2008) | Q | |
| | | 5000 | Kühne et al. (2005) | Q | |
| | $5.2\times10^{-2}$ | | Yaffe et al. (2003) | Q | 248, 249 |
| | $3.7\times10^{-2}$ | | Yao et al. (2002) | Q | 229, 267 |
| | $5.2\times10^{-2}$ | | English and Carroll (2001) | Q | 230, 274 |
| | $5.4\times10^{-2}$ | | Katritzky et al. (1998) | Q | |
| | $8.2\times10^{-2}$ | | Nirmalakhandan et al. (1997) | Q | |
| | | 5100 | Kühne et al. (2005) | ? | |
| | $5.7\times10^{-2}$ | | Yaws (1999) | ? | 21 |
| | $5.1\times10^{-2}$ | | Abraham et al. (1990) | ? | |
| 2-methylpropanedial | $6.5\times10^1$ | | Wang et al. (2017) | Q | 80, 238 |
| $C_4H_6O_2$ | $1.2\times10^1$ | | Wang et al. (2017) | Q | 80, 239 |
| [16002-19-0] | $1.9$ | | Wang et al. (2017) | Q | 80, 240 |
| VXYSFSCCSQAYJV-UHFFFAOYSA-N | | | | | |
| MCM:MALDIAL | $2.3\times10^2$ | | Wang et al. (2017) | Q | 80, 238 |
| $C_4H_4O_2$ | $4.6\times10^2$ | | Wang et al. (2017) | Q | 80, 239 |
| JGEMYUOFGVHXKV-UHFFFAOYSA-N | $6.5$ | | Wang et al. (2017) | Q | 80, 240 |
| MCM:C3MDIALOH | $1.4\times10^3$ | | Wang et al. (2017) | Q | 80, 238 |
| $C_4H_6O_3$ | $2.0\times10^3$ | | Wang et al. (2017) | Q | 80, 239 |
| SQHUBVCIVAIUAB-UHFFFAOYSA-N | $2.0$ | | Wang et al. (2017) | Q | 80, 240 |
| MCM:C41OH | $2.5\times10^7$ | | Wang et al. (2017) | Q | 80, 238 |
| $C_4H_8O_4$ | $3.7\times10^7$ | | Wang et al. (2017) | Q | 80, 239 |
| YTBSYETUWUMLBZ-UHFFFAOYSA-N | $1.2\times10^4$ | | Wang et al. (2017) | Q | 80, 240 |
| MCM:C41OOH | $3.3\times10^9$ | | Wang et al. (2017) | Q | 80, 238 |
| $C_4H_8O_5$ | $3.6\times10^8$ | | Wang et al. (2017) | Q | 80, 239 |
| ROHPNOOUQDXFHZ-UHFFFAOYSA-N | $1.4\times10^6$ | | Wang et al. (2017) | Q | 80, 240 |



Table A3.5: Aldehydes (RCHO) (...continued)

| Substance Formula (Trivial Name) [CAS Registry Number] InChIKey | $H_s^{cp}$ (at $T^\ominus$) $\left[\dfrac{\mathrm{mol}}{\mathrm{m^3\,Pa}}\right]$ | $\dfrac{\mathrm{d}\ln H_s^{cp}}{\mathrm{d}(1/T)}$ [K] | Reference | Type | Note |
|---|---|---|---|---|---|
| MCM:C4OCCOHCOH | $1.4\times10^4$ | | Wang et al. (2017) | Q | 80, 238 |
| $C_4H_8O_3$ | $5.0\times10^4$ | | Wang et al. (2017) | Q | 80, 239 |
| DFFAMJFPVBTTBX-UHFFFAOYSA-N | $3.5\times10^2$ | | Wang et al. (2017) | Q | 80, 240 |
| MCM:CHOC3DIOL | $2.6\times10^5$ | | Wang et al. (2017) | Q | 80, 238 |
| $C_4H_8O_3$ | $2.2\times10^5$ | | Wang et al. (2017) | Q | 80, 239 |
| CQSYGAZTCJHVFE-UHFFFAOYSA-N | $3.6\times10^3$ | | Wang et al. (2017) | Q | 80, 240 |
| MCM:HC3CCHO | $3.0\times10^1$ | | Wang et al. (2017) | Q | 80, 238 |
| $C_4H_6O_2$ | $8.7\times10^1$ | | Wang et al. (2017) | Q | 80, 239 |
| PPNVQCFSKPIRKK-UHFFFAOYSA-N | 1.7 | | Wang et al. (2017) | Q | 80, 240 |
| MCM:HC3CHO | $7.1\times10^2$ | | Wang et al. (2017) | Q | 80, 238 |
| $C_4H_6O_2$ | $3.7\times10^3$ | | Wang et al. (2017) | Q | 80, 239 |
| FXCMZPXXCRHRNK-UHFFFAOYSA-N | $2.5\times10^4$ | | Wang et al. (2017) | Q | 80, 240 |
| MCM:HMACROH | $1.5\times10^7$ | | Wang et al. (2017) | Q | 80, 238 |
| $C_4H_8O_4$ | $8.5\times10^7$ | | Wang et al. (2017) | Q | 80, 239 |
| OUEKYUKJSLEOIU-UHFFFAOYSA-N | $1.3\times10^4$ | | Wang et al. (2017) | Q | 80, 240 |
| MCM:HMACR | $4.1\times10^2$ | | Wang et al. (2017) | Q | 80, 238 |
| $C_4H_6O_2$ | $2.0\times10^2$ | | Wang et al. (2017) | Q | 80, 239 |
| QVBICLGJAQXLSA-UHFFFAOYSA-N | $1.1\times10^2$ | | Wang et al. (2017) | Q | 80, 240 |
| MCM:HO13C3CHO | $2.6\times10^4$ | | Wang et al. (2017) | Q | 80, 238 |
| $C_4H_8O_3$ | $2.3\times10^5$ | | Wang et al. (2017) | Q | 80, 239 |
| NKVLMFFGFYHDNE-UHFFFAOYSA-N | $3.2\times10^3$ | | Wang et al. (2017) | Q | 80, 240 |
| MCM:HO2C3CHO | $2.1\times10^2$ | | Wang et al. (2017) | Q | 80, 238 |
| $C_4H_8O_2$ | $1.4\times10^3$ | | Wang et al. (2017) | Q | 80, 239 |
| HSJKGGMUJITCBW-UHFFFAOYSA-N | $3.6\times10^1$ | | Wang et al. (2017) | Q | 80, 240 |
| MCM:HO3C3CHO | $1.1\times10^1$ | | Wang et al. (2017) | Q | 80, 238 |
| $C_4H_8O_2$ | $8.9\times10^1$ | | Wang et al. (2017) | Q | 80, 239 |
| UIKQNMXWCYQNCS-UHFFFAOYSA-N | 1.5 | | Wang et al. (2017) | Q | 80, 240 |
| MCM:HOC3H6CHO | $1.7\times10^2$ | | Wang et al. (2017) | Q | 80, 238 |
| $C_4H_8O_2$ | $2.8\times10^3$ | | Wang et al. (2017) | Q | 80, 239 |
| PIAOXUVIBAKVSP-UHFFFAOYSA-N | $6.5\times10^2$ | | Wang et al. (2017) | Q | 80, 240 |
| MCM:HOHOC4DIAL | $6.3\times10^5$ | | Wang et al. (2017) | Q | 80, 238 |
| $C_4H_6O_4$ | $3.0\times10^6$ | | Wang et al. (2017) | Q | 80, 239 |
| UUWVJXZLAWXQBU-UHFFFAOYSA-N | $2.8\times10^2$ | | Wang et al. (2017) | Q | 80, 240 |
| MCM:HOIPRCHO | $2.1\times10^2$ | | Wang et al. (2017) | Q | 80, 238 |
| $C_4H_8O_2$ | $6.8\times10^2$ | | Wang et al. (2017) | Q | 80, 239 |
| JTMCAHGCWBGWRV-UHFFFAOYSA-N | $2.5\times10^1$ | | Wang et al. (2017) | Q | 80, 240 |
| MCM:IBUTALOH | 6.6 | | Wang et al. (2017) | Q | 80, 238 |
| $C_4H_8O_2$ | $4.4\times10^1$ | | Wang et al. (2017) | Q | 80, 239 |
| HNVAGBIANFAIIL-UHFFFAOYSA-N | 1.5 | | Wang et al. (2017) | Q | 80, 240 |



Table A3.5: Aldehydes (RCHO) (...continued)

| Substance Formula (Trivial Name) [CAS Registry Number] InChIKey | $H_s^{cp}$ (at $T^{\ominus}$) $\left[\dfrac{\text{mol}}{\text{m}^3\,\text{Pa}}\right]$ | $\dfrac{\text{d}\ln H_s^{cp}}{\text{d}(1/T)}$ [K] | Reference | Type | Note |
|---|---|---|---|---|---|
| MCM:MACROH | $8.0\times10^3$ | | Wang et al. (2017) | Q | 80, 238 |
| $C_4H_8O_3$ | $3.6\times10^4$ | | Wang et al. (2017) | Q | 80, 239 |
| JBCPUXACCOWZEB-UHFFFAOYSA-N | $1.7\times10^2$ | | Wang et al. (2017) | Q | 80, 240 |
| pentanal | $6.6\times10^{-2}$ | 6500 | Brockbank (2013) | L | 1 |
| $C_4H_9CHO$ | $6.8\times10^{-2}$ | | Liu et al. (2015) | M | 72 |
| (valeraldehyde) | $3.9\times10^{-2}$ | | Kim and Kim (2014) | M | |
| [110-62-3] | $7.1\times10^{-2}$ | 6100 | Ji and Evans (2007) | M | |
| HGBOYTHUEUWSSQ-UHFFFAOYSA-N | $6.3\times10^{-2}$ | 6300 | Zhou and Mopper (1990) | M | 456 |
| | $6.7\times10^{-2}$ | | Buttery et al. (1969) | M | |
| | $5.8\times10^{-2}$ | | Buttery et al. (1965) | M | |
| | $2.4\times10^{-1}$ | 3800 | Djerki and Laub (1988) | V | |
| | $6.4\times10^{-2}$ | | Amoore and Buttery (1978) | V | |
| | $3.0\times10^{-2}$ | | Yaws (2003) | X | 258 |
| | $3.0\times10^{-2}$ | | Yaws (2003) | X | 237 |
| | $6.1\times10^{-2}$ | | Dupeux et al. (2022) | Q | 259 |
| | $2.2\times10^{-1}$ | | Keshavarz et al. (2022) | Q | |
| | $9.8\times10^{-2}$ | | Duchowicz et al. (2020) | Q | 299 |
| | $5.8\times10^{-2}$ | | Wang et al. (2017) | Q | 80, 238 |
| | $8.9\times10^{-2}$ | | Wang et al. (2017) | Q | 80, 239 |
| | $3.2\times10^{-2}$ | | Wang et al. (2017) | Q | 80, 240 |
| | $5.8\times10^{-3}$ | | Gharagheizi et al. (2012) | Q | |
| | $6.2\times10^{-2}$ | | Raventos-Duran et al. (2010) | Q | 242, 243 |
| | $7.8\times10^{-2}$ | | Raventos-Duran et al. (2010) | Q | 244 |
| | $6.2\times10^{-2}$ | | Raventos-Duran et al. (2010) | Q | 245 |
| | $4.1\times10^{-2}$ | | Gharagheizi et al. (2010) | Q | 246 |
| | $7.2\times10^{-2}$ | | Hilal et al. (2008) | Q | |
| | $1.2\times10^{-1}$ | | Modarresi et al. (2007) | Q | 67 |
| | | 6200 | Kühne et al. (2005) | Q | |
| | $6.7\times10^{-2}$ | | Yaffe et al. (2003) | Q | 248, 249 |
| | $2.9\times10^{-2}$ | | Yao et al. (2002) | Q | 229, 267 |
| | $6.7\times10^{-2}$ | | English and Carroll (2001) | Q | 230, 231 |
| | $7.3\times10^{-2}$ | | Nirmalakhandan et al. (1997) | Q | |
| | $6.1\times10^{-2}$ | | Suzuki et al. (1992) | Q | 232 |
| | $6.2\times10^{-2}$ | | Meylan and Howard (1991) | Q | |
| | $6.7\times10^{-2}$ | | Duchowicz et al. (2020) | ? | 185, 21 |
| | $6.7\times10^{-2}$ | | Mackay et al. (2006c) | ? | 21 |
| | | 5500 | Kühne et al. (2005) | ? | |
| | $3.0\times10^{-2}$ | | Yaws (1999) | ? | 21 |
| | $4.4\times10^{-2}$ | | Yaws and Yang (1992) | ? | 21, 38 |
| | $6.7\times10^{-2}$ | | Abraham et al. (1990) | ? | |
| 2-methylbutanal | $2.5\times10^{-2}$ | 5600 | Brockbank (2013) | L | 1 |
| $C_5H_{10}O$ | $2.3\times10^{-2}$ | | Pollien et al. (2003) | M | |
| [96-17-3] | $5.9\times10^{-2}$ | | Wang et al. (2017) | Q | 80, 238 |
| BYGQBDHUGHBGMD-UHFFFAOYSA-N | $8.7\times10^{-2}$ | | Wang et al. (2017) | Q | 80, 239 |
| | $3.9\times10^{-2}$ | | Wang et al. (2017) | Q | 80, 240 |
| | $9.5\times10^{-3}$ | | Hertel et al. (2007) | Q | 467 |





Table A3.5: Aldehydes (RCHO) (...continued)

| Substance<br>Formula<br>(Trivial Name)<br>[CAS Registry Number]<br>InChIKey | $H_s^{cp}$<br>(at $T^\ominus$)<br>$\left[\dfrac{\text{mol}}{\text{m}^3\,\text{Pa}}\right]$ | $\dfrac{\mathrm{d}\ln H_s^{cp}}{\mathrm{d}(1/T)}$<br><br>[K] | Reference | Type | Note |
|---|---|---|---|---|---|
| 3-methylbutanal | $1.1\times10^{-2}$ | 10000 | Bruneel et al. (2016) | M | 33 |
| $C_5H_{10}O$ | $2.4\times10^{-2}$ | 6100 | Wieland et al. (2015) | M | 468 |
| (isovaleraldehyde) | $2.1\times10^{-2}$ | | Kim and Kim (2014) | M | |
| [590-86-3] | $2.6\times10^{-2}$ | | Pollien et al. (2003) | M | |
| YGHRJJRRZDOVPD-UHFFFAOYSA-N | $2.0\times10^{-2}$ | | Nelson and Hoff (1968) | M | 297 |
| | $2.4\times10^{-2}$ | | Duchowicz et al. (2020) | V | 186 |
| | $2.5\times10^{-2}$ | | HSDB (2015) | V | |
| | $5.2\times10^{-2}$ | | Yaws (2003) | X | 237 |
| | $3.8\times10^{-2}$ | | Duchowicz et al. (2020) | Q | |
| | $5.9\times10^{-2}$ | | Wang et al. (2017) | Q | 80, 238 |
| | $1.1\times10^{-1}$ | | Wang et al. (2017) | Q | 80, 239 |
| | $4.5\times10^{-2}$ | | Wang et al. (2017) | Q | 80, 240 |
| | $5.5\times10^{-2}$ | | Gharagheizi et al. (2010) | Q | 246 |
| | $7.3\times10^{-2}$ | | Hilal et al. (2008) | Q | |
| | $9.1\times10^{-2}$ | | Modarresi et al. (2007) | Q | 67 |
| | $9.8\times10^{-3}$ | | Hertel et al. (2007) | Q | 467 |
| MCM:C4MDIAL | $1.6\times10^{2}$ | | Wang et al. (2017) | Q | 80, 238 |
| $C_5H_6O_2$ | $6.5\times10^{2}$ | | Wang et al. (2017) | Q | 80, 239 |
| USBJWIKCHJDWPF-UHFFFAOYSA-N | 2.8 | | Wang et al. (2017) | Q | 80, 240 |
| MCM:CO1M22CHO | $3.7\times10^{1}$ | | Wang et al. (2017) | Q | 80, 238 |
| $C_5H_8O_2$ | 8.9 | | Wang et al. (2017) | Q | 80, 239 |
| WNBFTLCNQKKVHC-UHFFFAOYSA-N | $3.4\times10^{-1}$ | | Wang et al. (2017) | Q | 80, 240 |
| 2,2-dimethylpropanal | $4.1\times10^{-2}$ | | Wang et al. (2017) | Q | 80, 238 |
| $C_5H_{10}O$ | $5.6\times10^{-2}$ | | Wang et al. (2017) | Q | 80, 239 |
| (pivaldehyde) | $2.8\times10^{-2}$ | | Wang et al. (2017) | Q | 80, 240 |
| [630-19-3] | | | | | |
| FJJYHTVHBVXEEQ-UHFFFAOYSA-N | | | | | |
| MCM:C3EDIALOH | $1.0\times10^{3}$ | | Wang et al. (2017) | Q | 80, 238 |
| $C_5H_8O_3$ | $2.6\times10^{3}$ | | Wang et al. (2017) | Q | 80, 239 |
| QTDYRJFEJDMBDI-UHFFFAOYSA-N | 1.1 | | Wang et al. (2017) | Q | 80, 240 |
| MCM:C3M3OH2CHO | $1.0\times10^{1}$ | | Wang et al. (2017) | Q | 80, 238 |
| $C_5H_{10}O_2$ | $6.5\times10^{1}$ | | Wang et al. (2017) | Q | 80, 239 |
| VKYKDJZVZBURQF-UHFFFAOYSA-N | $6.8\times10^{-1}$ | | Wang et al. (2017) | Q | 80, 240 |
| MCM:C42CHO | $2.5\times10^{4}$ | | Wang et al. (2017) | Q | 80, 238 |
| $C_5H_{10}O_3$ | $3.2\times10^{5}$ | | Wang et al. (2017) | Q | 80, 239 |
| NGVSSTUGOBBIBC-UHFFFAOYSA-N | $1.4\times10^{3}$ | | Wang et al. (2017) | Q | 80, 240 |
| MCM:C4M2AL2OH | $3.5\times10^{5}$ | | Wang et al. (2017) | Q | 80, 238 |
| $C_5H_8O_4$ | $5.0\times10^{6}$ | | Wang et al. (2017) | Q | 80, 239 |
| NUHRTSFMSDZSTE-UHFFFAOYSA-N | $1.0\times10^{2}$ | | Wang et al. (2017) | Q | 80, 240 |
| MCM:C4OHCHO | 8.3 | | Wang et al. (2017) | Q | 80, 238 |
| $C_5H_{10}O_2$ | $5.6\times10^{1}$ | | Wang et al. (2017) | Q | 80, 239 |
| SUTLBTHMXYSMSZ-UHFFFAOYSA-N | 1.3 | | Wang et al. (2017) | Q | 80, 240 |



Table A3.5: Aldehydes (RCHO) (...continued)

| Substance Formula (Trivial Name) [CAS Registry Number] InChIKey | $H_s^{cp}$ (at $T^\ominus$) $\left[\dfrac{\mathrm{mol}}{\mathrm{m}^3\,\mathrm{Pa}}\right]$ | $\dfrac{\mathrm{d}\ln H_s^{cp}}{\mathrm{d}(1/T)}$ [K] | Reference | Type | Note |
|---|---|---|---|---|---|
| MCM:C514OH | $1.5\times10^5$ | | Wang et al. (2017) | Q | 80, 238 |
| $C_5H_8O_3$ | $8.7\times10^5$ | | Wang et al. (2017) | Q | 80, 239 |
| UIVALZXFWDRKKE-UHFFFAOYSA-N | $1.8\times10^3$ | | Wang et al. (2017) | Q | 80, 240 |
| MCM:C57OH | $1.4\times10^7$ | | Wang et al. (2017) | Q | 80, 238 |
| $C_5H_{10}O_4$ | $6.8\times10^7$ | | Wang et al. (2017) | Q | 80, 239 |
| ZKDSJDSEVDWBAC-UHFFFAOYSA-N | $1.9\times10^3$ | | Wang et al. (2017) | Q | 80, 240 |
| MCM:C58OH | $1.4\times10^7$ | | Wang et al. (2017) | Q | 80, 238 |
| $C_5H_{10}O_4$ | $1.4\times10^8$ | | Wang et al. (2017) | Q | 80, 239 |
| HTPZSALIZDTBIL-UHFFFAOYSA-N | $2.9\times10^3$ | | Wang et al. (2017) | Q | 80, 240 |
| MCM:C5DIALOH | $3.1\times10^4$ | | Wang et al. (2017) | Q | 80, 238 |
| $C_5H_6O_3$ | $5.8\times10^4$ | | Wang et al. (2017) | Q | 80, 239 |
| KBKQEUDBRUGUIQ-UHFFFAOYSA-N | $3.3\times10^3$ | | Wang et al. (2017) | Q | 80, 240 |
| MCM:H13C43CHO | $1.4\times10^4$ | | Wang et al. (2017) | Q | 80, 238 |
| $C_5H_{10}O_3$ | $1.7\times10^5$ | | Wang et al. (2017) | Q | 80, 239 |
| KJLUILOCNALPNM-UHFFFAOYSA-N | $2.0\times10^3$ | | Wang et al. (2017) | Q | 80, 240 |
| MCM:H2M2C3CHO | $1.1\times10^2$ | | Wang et al. (2017) | Q | 80, 238 |
| $C_5H_{10}O_2$ | $6.2\times10^2$ | | Wang et al. (2017) | Q | 80, 239 |
| FXFBPKDQLDOIRG-UHFFFAOYSA-N | $2.2\times10^1$ | | Wang et al. (2017) | Q | 80, 240 |
| MCM:HC4ACHO | $4.8\times10^2$ | | Wang et al. (2017) | Q | 80, 238 |
| $C_5H_8O_2$ | $4.6\times10^3$ | | Wang et al. (2017) | Q | 80, 239 |
| BSHDRMLUCYMQOP-UHFFFAOYSA-N | $1.7\times10^4$ | | Wang et al. (2017) | Q | 80, 240 |
| MCM:HC4CCHO | $1.2\times10^3$ | 11000 | Wieser et al. (2023) | Q | 437 |
| $C_5H_8O_2$ | $4.8\times10^2$ | | Wang et al. (2017) | Q | 80, 238 |
| GCHJBJOOADXJFT-UHFFFAOYSA-N | $4.1\times10^3$ | | Wang et al. (2017) | Q | 80, 239 |
| | $5.6\times10^3$ | | Wang et al. (2017) | Q | 80, 240 |
| MCM:HM22CHO | $1.1\times10^2$ | | Wang et al. (2017) | Q | 80, 238 |
| $C_5H_{10}O_2$ | $3.7\times10^2$ | | Wang et al. (2017) | Q | 80, 239 |
| JJMOMMLADQPZNY-UHFFFAOYSA-N | $1.1\times10^1$ | | Wang et al. (2017) | Q | 80, 240 |
| MCM:HO24C4CHO | $2.5\times10^4$ | | Wang et al. (2017) | Q | 80, 238 |
| $C_5H_{10}O_3$ | $3.6\times10^5$ | | Wang et al. (2017) | Q | 80, 239 |
| WDKLWOBHKQJYSU-UHFFFAOYSA-N | $1.2\times10^3$ | | Wang et al. (2017) | Q | 80, 240 |
| MCM:HO2C43CHO | $2.0\times10^2$ | | Wang et al. (2017) | Q | 80, 238 |
| $C_5H_{10}O_2$ | $6.9\times10^2$ | | Wang et al. (2017) | Q | 80, 239 |
| ZURZPPULRFXVLF-UHFFFAOYSA-N | $3.3\times10^1$ | | Wang et al. (2017) | Q | 80, 240 |
| MCM:HO2C4CHO | $1.6\times10^2$ | | Wang et al. (2017) | Q | 80, 238 |
| $C_5H_{10}O_2$ | $2.1\times10^3$ | | Wang et al. (2017) | Q | 80, 239 |
| HFZMJAMTNAAZQE-UHFFFAOYSA-N | $4.1\times10^2$ | | Wang et al. (2017) | Q | 80, 240 |
| MCM:HO3C4CHO | $1.6\times10^2$ | | Wang et al. (2017) | Q | 80, 238 |
| $C_5H_{10}O_2$ | $7.8\times10^2$ | | Wang et al. (2017) | Q | 80, 239 |
| WRWLNLBWBJEUPI-UHFFFAOYSA-N | $1.6\times10^1$ | | Wang et al. (2017) | Q | 80, 240 |





Table A3.5: Aldehydes (RCHO) (...continued)

| Substance Formula (Trivial Name) [CAS Registry Number] InChIKey | $H_s^{cp}$ (at $T^\ominus$) $\left[\dfrac{\text{mol}}{\text{m}^3\,\text{Pa}}\right]$ | $\dfrac{\text{d}\ln H_s^{cp}}{\text{d}(1/T)}$ [K] | Reference | Type | Note |
|---|---|---|---|---|---|
| MCM:HOBUT2CHO | $1.6\times10^2$ | | Wang et al. (2017) | Q | 80, 238 |
| $C_5H_{10}O_2$ | $1.6\times10^3$ | | Wang et al. (2017) | Q | 80, 239 |
| PLBZJQQQFIOXRU-UHFFFAOYSA-N | $2.4\times10^2$ | | Wang et al. (2017) | Q | 80, 240 |
| MCM:HOIBUTCHO | $1.6\times10^2$ | | Wang et al. (2017) | Q | 80, 238 |
| $C_5H_{10}O_2$ | $2.0\times10^3$ | | Wang et al. (2017) | Q | 80, 239 |
| RWXBAXNFTXQJAY-UHFFFAOYSA-N | $4.8\times10^2$ | | Wang et al. (2017) | Q | 80, 240 |
| MCM:MBOBCO | $7.4\times10^3$ | | Wang et al. (2017) | Q | 80, 238 |
| $C_5H_{10}O_3$ | $4.4\times10^4$ | | Wang et al. (2017) | Q | 80, 239 |
| YDXYYBJRCIQQSF-UHFFFAOYSA-N | $1.5\times10^2$ | | Wang et al. (2017) | Q | 80, 240 |
| MCM:PROL1MCHO | $5.8$ | | Wang et al. (2017) | Q | 80, 238 |
| $C_5H_{10}O_2$ | $4.2\times10^1$ | | Wang et al. (2017) | Q | 80, 239 |
| MEHIGMLPKIJWEA-UHFFFAOYSA-N | $7.4\times10^{-1}$ | | Wang et al. (2017) | Q | 80, 240 |
| hexanal | $4.5\times10^{-2}$ | 6400 | Brockbank (2013) | L | 1 |
| $C_5H_{11}CHO$ | $2.3\times10^{-2}$ | 5200 | Kutsuna and Kaneyasu (2021) | M | |
| [66-25-1] | $2.9\times10^{-2}$ | 8900 | Bruneel et al. (2016) | M | |
| JARKCYVAAOWBJS-UHFFFAOYSA-N | $4.7\times10^{-2}$ | | Souchon et al. (2004) | M | |
| | $3.2\times10^{-2}$ | | Karl et al. (2003) | M | |
| | $1.6\times10^{-1}$ | 4900 | Meynier et al. (2003) | M | 38 |
| | $2.7\times10^{-2}$ | | van Ruth et al. (2002) | M | 14 |
| | $2.2\times10^{-2}$ | | van Ruth and Villeneuve (2002) | M | 14, 361 |
| | $1.6\times10^{-2}$ | | van Ruth et al. (2001) | M | 14 |
| | $4.9\times10^{-2}$ | 6500 | Zhou and Mopper (1990) | M | 456 |
| | $4.6\times10^{-2}$ | | Buttery et al. (1969) | M | |
| | $5.8\times10^{-2}$ | | Buttery et al. (1965) | M | |
| | $3.5\times10^{-2}$ | | Amoore and Buttery (1978) | V | |
| | $2.8\times10^{-2}$ | | Yaws (2003) | X | 258 |
| | $2.8\times10^{-2}$ | | Yaws (2003) | X | 237, 38 |
| | $4.8\times10^{-2}$ | | Sieg et al. (2008) | C | |
| | $4.6\times10^{-2}$ | | Meynier et al. (2003) | C | |
| | $3.1\times10^{-2}$ | | Nahon et al. (2000) | C | 14 |
| | $4.2\times10^{-2}$ | | Dupeux et al. (2022) | Q | 259 |
| | $3.0\times10^{-1}$ | | Keshavarz et al. (2022) | Q | |
| | $9.8\times10^{-2}$ | | Duchowicz et al. (2020) | Q | 299 |
| | $4.6\times10^{-2}$ | | Wang et al. (2017) | Q | 80, 238 |
| | $5.8\times10^{-2}$ | | Wang et al. (2017) | Q | 80, 239 |
| | $5.5\times10^{-2}$ | | Wang et al. (2017) | Q | 80, 240 |
| | $4.6\times10^{-2}$ | | Li et al. (2014) | Q | 241 |
| | $4.3\times10^{-3}$ | | Gharagheizi et al. (2012) | Q | |
| | $4.9\times10^{-2}$ | | Raventos-Duran et al. (2010) | Q | 242, 243 |
| | $4.9\times10^{-2}$ | | Raventos-Duran et al. (2010) | Q | 244 |
| | $4.9\times10^{-2}$ | | Raventos-Duran et al. (2010) | Q | 245 |
| | $2.9\times10^{-2}$ | | Gharagheizi et al. (2010) | Q | 246 |
| | $5.8\times10^{-2}$ | | Hilal et al. (2008) | Q | |
| | $9.8\times10^{-2}$ | | Modarresi et al. (2007) | Q | 67 |
| | $1.1\times10^{-2}$ | | Hertel et al. (2007) | Q | 467 |





Table A3.5: Aldehydes (RCHO) (...continued)

| Substance / Formula / (Trivial Name) / [CAS Registry Number] / InChIKey | $H_s^{cp}$ (at $T^{\ominus}$) $\left[\dfrac{\mathrm{mol}}{\mathrm{m^3\,Pa}}\right]$ | $\dfrac{\mathrm{d}\ln H_s^{cp}}{\mathrm{d}(1/T)}$ [K] | Reference | Type | Note |
|---|---|---|---|---|---|
| | | 6600 | Kühne et al. (2005) | Q | |
| | $4.6\times10^{-2}$ | | Yaffe et al. (2003) | Q | 248, 249 |
| | $2.6\times10^{-2}$ | | Yao et al. (2002) | Q | 229 |
| | $5.2\times10^{-2}$ | | English and Carroll (2001) | Q | 230, 260 |
| | $4.8\times10^{-2}$ | | Katritzky et al. (1998) | Q | |
| | $5.8\times10^{-2}$ | | Nirmalakhandan et al. (1997) | Q | |
| | $4.6\times10^{-2}$ | | Suzuki et al. (1992) | Q | 232 |
| | $4.6\times10^{-2}$ | | Duchowicz et al. (2020) | ? | 185, 21 |
| | $4.6\times10^{-2}$ | | Mackay et al. (2006c) | ? | 21 |
| | | 6900 | Kühne et al. (2005) | ? | |
| | $3.8\times10^{-2}$ | | Yaws (1999) | ? | 21, 38 |
| | $1.9\times10^{-2}$ | | Yaws and Yang (1992) | ? | 21, 38 |
| | $4.6\times10^{-2}$ | | Abraham et al. (1990) | ? | |
| pentanedial OHC(CH$_2$)$_3$CHO (glutaraldehyde) [111-30-8] SXRSQZLOMIGNAQ-UHFFFAOYSA-N | $3.0\times10^{2}$ $4.1\times10^{2}$ | 9200 8800 9100 | Olson (1998) HSDB (2015) Kühne et al. (2005) Kühne et al. (2005) | M Q Q ? | 99 |
| 2-methylpentanal C$_6$H$_{12}$O (2-methylvaleraldehyde) [123-15-9] FTZILAQGHINQQR-UHFFFAOYSA-N | $5.3\times10^{-2}$ $5.6\times10^{-2}$ $3.5\times10^{-2}$ $2.7\times10^{-2}$ | 5700 5300 | Wang et al. (2017) Wang et al. (2017) Wang et al. (2017) HSDB (2015) Kühne et al. (2005) Kühne et al. (2005) | Q Q Q Q Q ? | 80, 238 80, 239 80, 240 99 |
| hexanedial C$_6$H$_{10}$O$_2$ (adipaldehyde) [1072-21-5] UMHJEEQLYBKSAN-UHFFFAOYSA-N | $4.2\times10^{1}$ $1.7\times10^{2}$ $3.2\times10^{1}$ | | Wang et al. (2017) Wang et al. (2017) Wang et al. (2017) | Q Q Q | 80, 238 80, 239 80, 240 |
| MCM:M22C3CHO C$_6$H$_{12}$O LTNUSYNQZJZUSY-UHFFFAOYSA-N | $3.2\times10^{-2}$ $6.5\times10^{-2}$ $3.1\times10^{-2}$ | | Wang et al. (2017) Wang et al. (2017) Wang et al. (2017) | Q Q Q | 80, 238 80, 239 80, 240 |
| MCM:M2C43CHO C$_6$H$_{12}$O AKUUEDVRXOZTBF-UHFFFAOYSA-N | $6.2\times10^{-2}$ $6.0\times10^{-2}$ $3.7\times10^{-2}$ | | Wang et al. (2017) Wang et al. (2017) Wang et al. (2017) | Q Q Q | 80, 238 80, 239 80, 240 |
| MCM:M33C3CHO C$_6$H$_{12}$O QYPLKDUOPJZROX-UHFFFAOYSA-N | $3.2\times10^{-2}$ $5.0\times10^{-2}$ $2.6\times10^{-2}$ | | Wang et al. (2017) Wang et al. (2017) Wang et al. (2017) | Q Q Q | 80, 238 80, 239 80, 240 |
| 3-methylpentanal C$_6$H$_{12}$O [15877-57-3] YJWJGLQYQJGEEP-UHFFFAOYSA-N | $3.8\times10^{-2}$ $5.3\times10^{-2}$ $7.8\times10^{-2}$ $4.5\times10^{-2}$ $3.8\times10^{-2}$ | | Yaws (2003) Wang et al. (2017) Wang et al. (2017) Wang et al. (2017) Gharagheizi et al. (2010) | X Q Q Q Q | 237 80, 238 80, 239 80, 240 246 |



Table A3.5: Aldehydes (RCHO) (. . . continued)

| Substance<br>Formula<br>(Trivial Name)<br>[CAS Registry Number]<br>InChIKey | $H_s^{cp}$<br>(at $T^\ominus$)<br>$\left[\dfrac{\text{mol}}{\text{m}^3\,\text{Pa}}\right]$ | $\dfrac{\mathrm{d}\ln H_s^{cp}}{\mathrm{d}(1/T)}$<br><br>[K] | Reference | Type | Note |
|---|---|---|---|---|---|
| 4-methylpentanal<br>$C_6H_{12}O$<br>[1119-16-0]<br>JGEGJYXHCFUMJF-UHFFFAOYSA-N | $3.8\times10^{-2}$<br>$3.8\times10^{-2}$ | | Yaws (2003)<br>Gharagheizi et al. (2010) | X<br>Q | 237<br>246 |
| 2-ethylbutanal<br>$C_6H_{12}O$<br>[97-96-1]<br>UNNGUFMVYQJGTD-UHFFFAOYSA-N | $4.0\times10^{-2}$<br>$3.7\times10^{-2}$ | | Yaws (2003)<br>Dupeux et al. (2022) | X<br>Q | 258<br>259 |
| MCM:C518CHO<br>$C_6H_{10}O_2$<br>XEAYIEUKFACAKS-UHFFFAOYSA-N | $3.2\times10^{2}$<br>$5.3\times10^{2}$<br>$2.8\times10^{1}$ | | Wang et al. (2017)<br>Wang et al. (2017)<br>Wang et al. (2017) | Q<br>Q<br>Q | 80, 238<br>80, 239<br>80, 240 |
| MCM:C615OH<br>$C_6H_{10}O_3$<br>LOUHJSYYTNSRBN-UHFFFAOYSA-N | $4.9\times10^{3}$<br>$4.4\times10^{4}$<br>$3.7\times10^{1}$ | | Wang et al. (2017)<br>Wang et al. (2017)<br>Wang et al. (2017) | Q<br>Q<br>Q | 80, 238<br>80, 239<br>80, 240 |
| MCM:C623OH<br>$C_6H_{12}O_4$<br>ASGSKBSLHTVURJ-UHFFFAOYSA-N | $3.2\times10^{8}$<br>$6.2\times10^{8}$<br>$1.9\times10^{4}$ | | Wang et al. (2017)<br>Wang et al. (2017)<br>Wang et al. (2017) | Q<br>Q<br>Q | 80, 238<br>80, 239<br>80, 240 |
| MCM:C67OH<br>$C_6H_{12}O_3$<br>SIBVVDHXFPINCX-UHFFFAOYSA-N | $1.3\times10^{5}$<br>$3.5\times10^{5}$<br>$4.6\times10^{2}$ | | Wang et al. (2017)<br>Wang et al. (2017)<br>Wang et al. (2017) | Q<br>Q<br>Q | 80, 238<br>80, 239<br>80, 240 |
| MCM:C68OH<br>$C_6H_{12}O_3$<br>RJKLRJTWLKKQSE-UHFFFAOYSA-N | $1.3\times10^{5}$<br>$3.6\times10^{5}$<br>$3.6\times10^{2}$ | | Wang et al. (2017)<br>Wang et al. (2017)<br>Wang et al. (2017) | Q<br>Q<br>Q | 80, 238<br>80, 239<br>80, 240 |
| MCM:C6DIALOH<br>$C_6H_{10}O_3$<br>GATFIJYWERJMGU-UHFFFAOYSA-N | $1.2\times10^{5}$<br>$4.7\times10^{5}$<br>$2.6\times10^{3}$ | | Wang et al. (2017)<br>Wang et al. (2017)<br>Wang et al. (2017) | Q<br>Q<br>Q | 80, 238<br>80, 239<br>80, 240 |
| MCM:CO1C6OH<br>$C_6H_{12}O_2$<br>FPFTWHJPEMPAGE-UHFFFAOYSA-N | $1.3\times10^{2}$<br>$1.9\times10^{3}$<br>$1.4\times10^{3}$ | | Wang et al. (2017)<br>Wang et al. (2017)<br>Wang et al. (2017) | Q<br>Q<br>Q | 80, 238<br>80, 239<br>80, 240 |
| MCM:CO1H63OH<br>$C_6H_{12}O_3$<br>TWSXDPXQZVBPJU-UHFFFAOYSA-N | $4.1\times10^{5}$<br>$2.5\times10^{6}$<br>$2.8\times10^{4}$ | | Wang et al. (2017)<br>Wang et al. (2017)<br>Wang et al. (2017) | Q<br>Q<br>Q | 80, 238<br>80, 239<br>80, 240 |
| MCM:H2M2C4CHO<br>$C_6H_{12}O_2$<br>NEJZWIVQOAKZHY-UHFFFAOYSA-N | $9.3\times10^{1}$<br>$1.3\times10^{3}$<br>$2.0\times10^{2}$ | | Wang et al. (2017)<br>Wang et al. (2017)<br>Wang et al. (2017) | Q<br>Q<br>Q | 80, 238<br>80, 239<br>80, 240 |
| MCM:H2M3C4CHO<br>$C_6H_{12}O_2$<br>GUXDGBBNSMURMG-UHFFFAOYSA-N | $1.5\times10^{2}$<br>$1.8\times10^{3}$<br>$1.5\times10^{2}$ | | Wang et al. (2017)<br>Wang et al. (2017)<br>Wang et al. (2017) | Q<br>Q<br>Q | 80, 238<br>80, 239<br>80, 240 |
| MCM:H3M3C4CHO<br>$C_6H_{12}O_2$<br>HKXRCBUXVFAKDA-UHFFFAOYSA-N | $9.3\times10^{1}$<br>$5.5\times10^{2}$<br>$1.2\times10^{1}$ | | Wang et al. (2017)<br>Wang et al. (2017)<br>Wang et al. (2017) | Q<br>Q<br>Q | 80, 238<br>80, 239<br>80, 240 |



Table A3.5: Aldehydes (RCHO) (...continued)

| Substance Formula (Trivial Name) [CAS Registry Number] InChIKey | $H_s^{cp}$ (at $T^\ominus$) $\left[\dfrac{\mathrm{mol}}{\mathrm{m^3\,Pa}}\right]$ | $\dfrac{\mathrm{d}\ln H_s^{cp}}{\mathrm{d}(1/T)}$ [K] | Reference | Type | Note |
|---|---|---|---|---|---|
| MCM:HM22C3CHO | $9.3\times10^1$ | | Wang et al. (2017) | Q | 80, 238 |
| $C_6H_{12}O_2$ | $1.1\times10^3$ | | Wang et al. (2017) | Q | 80, 239 |
| VOISQCUJWIZDHK-UHFFFAOYSA-N | $5.9\times10^1$ | | Wang et al. (2017) | Q | 80, 240 |
| MCM:HM2C43CHO | $1.5\times10^2$ | | Wang et al. (2017) | Q | 80, 238 |
| $C_6H_{12}O_2$ | $1.7\times10^3$ | | Wang et al. (2017) | Q | 80, 239 |
| QQZFDCJFBSAIBQ-UHFFFAOYSA-N | $1.6\times10^2$ | | Wang et al. (2017) | Q | 80, 240 |
| MCM:HM33C3CHO | $9.3\times10^1$ | | Wang et al. (2017) | Q | 80, 238 |
| $C_6H_{12}O_2$ | $1.0\times10^3$ | | Wang et al. (2017) | Q | 80, 239 |
| YHMMFPKTSXQULG-UHFFFAOYSA-N | $2.6\times10^2$ | | Wang et al. (2017) | Q | 80, 240 |
| MCM:HO2C54CHO | $1.5\times10^2$ | | Wang et al. (2017) | Q | 80, 238 |
| $C_6H_{12}O_2$ | $1.3\times10^3$ | | Wang et al. (2017) | Q | 80, 239 |
| ZHZOUKIXPREHLS-UHFFFAOYSA-N | $1.3\times10^2$ | | Wang et al. (2017) | Q | 80, 240 |
| MCM:HO3C5CHO | $1.4\times10^2$ | | Wang et al. (2017) | Q | 80, 238 |
| $C_6H_{12}O_2$ | $1.5\times10^3$ | | Wang et al. (2017) | Q | 80, 239 |
| HAPLZPWSLJPZBM-UHFFFAOYSA-N | $1.2\times10^2$ | | Wang et al. (2017) | Q | 80, 240 |
| MCM:HO5C5CHO | $6.9$ | | Wang et al. (2017) | Q | 80, 238 |
| $C_6H_{12}O_2$ | $4.0\times10^1$ | | Wang et al. (2017) | Q | 80, 239 |
| BRZMRZVKWQWYPJ-UHFFFAOYSA-N | $1.0$ | | Wang et al. (2017) | Q | 80, 240 |
| heptanal | $3.4\times10^{-2}$ | 7500 | Brockbank (2013) | L | 1 |
| $C_6H_{13}CHO$ | $3.5\times10^{-2}$ | | Souchon et al. (2004) | M | |
| [111-71-7] | $2.0\times10^{-2}$ | | van Ruth et al. (2002) | M | 14 |
| FXHGMKSSBGDXIY-UHFFFAOYSA-N | $1.9\times10^{-2}$ | | van Ruth and Villeneuve (2002) | M | 14, 361 |
| | $1.1\times10^{-2}$ | | van Ruth et al. (2001) | M | 14 |
| | $3.3\times10^{-2}$ | 7500 | Zhou and Mopper (1990) | M | 456 |
| | $3.7\times10^{-2}$ | | Buttery et al. (1969) | M | |
| | $6.0\times10^{-2}$ | | Buttery et al. (1965) | M | |
| | $5.4\times10^{-2}$ | | Amoore and Buttery (1978) | V | |
| | $2.1\times10^{-2}$ | | Yaws (2003) | X | 258 |
| | $2.0\times10^{-2}$ | | Yaws (2003) | X | 237, 38 |
| | $3.7\times10^{-2}$ | | Sieg et al. (2008) | C | |
| | $3.0\times10^{-2}$ | | Dupeux et al. (2022) | Q | 259 |
| | $5.3\times10^{-2}$ | | Keshavarz et al. (2022) | Q | |
| | $9.8\times10^{-2}$ | | Duchowicz et al. (2020) | Q | 184 |
| | $3.6\times10^{-2}$ | | Wang et al. (2017) | Q | 80, 238 |
| | $4.3\times10^{-2}$ | | Wang et al. (2017) | Q | 80, 239 |
| | $2.5\times10^{-2}$ | | Wang et al. (2017) | Q | 80, 240 |
| | $3.7\times10^{-2}$ | | Li et al. (2014) | Q | 241 |
| | $3.3\times10^{-3}$ | | Gharagheizi et al. (2012) | Q | |
| | $3.9\times10^{-2}$ | | Raventos-Duran et al. (2010) | Q | 242, 243 |
| | $3.9\times10^{-2}$ | | Raventos-Duran et al. (2010) | Q | 244 |
| | $3.9\times10^{-2}$ | | Raventos-Duran et al. (2010) | Q | 245 |
| | $2.1\times10^{-2}$ | | Gharagheizi et al. (2010) | Q | 246 |
| | $3.7\times10^{-2}$ | | Hilal et al. (2008) | Q | |
| | $8.1\times10^{-2}$ | | Modarresi et al. (2007) | Q | 67 |





Table A3.5: Aldehydes (RCHO) (…continued)

| Substance / Formula / (Trivial Name) / [CAS Registry Number] / InChIKey | $H_s^{cp}$ (at $T^\ominus$) $\left[\dfrac{\text{mol}}{\text{m}^3\,\text{Pa}}\right]$ | $\dfrac{\text{d}\ln H_s^{cp}}{\text{d}(1/T)}$ [K] | Reference | Type | Note |
|---|---|---|---|---|---|
| | | 6900 | Kühne et al. (2005) | Q | |
| | $3.7\times10^{-2}$ | | Yaffe et al. (2003) | Q | 248, 249 |
| | $2.2\times10^{-2}$ | | Yao et al. (2002) | Q | 229 |
| | $3.9\times10^{-2}$ | | English and Carroll (2001) | Q | 230, 274 |
| | $4.7\times10^{-2}$ | | Katritzky et al. (1998) | Q | |
| | $4.5\times10^{-2}$ | | Nirmalakhandan et al. (1997) | Q | |
| | $3.6\times10^{-2}$ | | Suzuki et al. (1992) | Q | 232 |
| | $3.7\times10^{-2}$ | | Duchowicz et al. (2020) | ? | 185, 21 |
| | | 7100 | Kühne et al. (2005) | ? | |
| | $2.8\times10^{-2}$ | | Yaws (1999) | ? | 21, 38 |
| | $2.3\times10^{-2}$ | | Yaws and Yang (1992) | ? | 21, 38 |
| | $3.7\times10^{-2}$ | | Abraham et al. (1990) | ? | |
| MCM:C622CHO $C_7H_{12}O_2$ JWVBPJJNIHELFZ-UHFFFAOYSA-N | $5.8\times10^{3}$ | 11000 | Wieser et al. (2023) | Q | 437 |
| | $2.5\times10^{2}$ | | Wang et al. (2017) | Q | 80, 238 |
| | $1.6\times10^{3}$ | | Wang et al. (2017) | Q | 80, 239 |
| | $7.6\times10^{1}$ | | Wang et al. (2017) | Q | 80, 240 |
| MCM:C624CHO $C_7H_{12}O_2$ AIMYSSDKEKDSEU-UHFFFAOYSA-N | $5.8\times10^{3}$ | 11000 | Wieser et al. (2023) | Q | 437 |
| | $2.5\times10^{2}$ | | Wang et al. (2017) | Q | 80, 238 |
| | $1.2\times10^{3}$ | | Wang et al. (2017) | Q | 80, 239 |
| | $2.7\times10^{2}$ | | Wang et al. (2017) | Q | 80, 240 |
| MCM:C728OH $C_7H_{14}O_4$ RQDWEXZQVQBWBC-UHFFFAOYSA-N | $2.9\times10^{8}$ | | Wang et al. (2017) | Q | 80, 238 |
| | $1.2\times10^{9}$ | | Wang et al. (2017) | Q | 80, 239 |
| | $2.2\times10^{6}$ | | Wang et al. (2017) | Q | 80, 240 |
| MCM:C730OH $C_7H_{14}O_4$ ROYLBPWMPALPRM-UHFFFAOYSA-N | $2.9\times10^{8}$ | | Wang et al. (2017) | Q | 80, 238 |
| | $3.4\times10^{9}$ | | Wang et al. (2017) | Q | 80, 239 |
| | $1.3\times10^{6}$ | | Wang et al. (2017) | Q | 80, 240 |
| MCM:H3M3C5CHO $C_7H_{14}O_2$ PUMVOOYIPAYIQQ-UHFFFAOYSA-N | $7.3\times10^{1}$ | | Wang et al. (2017) | Q | 80, 238 |
| | $1.1\times10^{3}$ | | Wang et al. (2017) | Q | 80, 239 |
| | $5.6\times10^{1}$ | | Wang et al. (2017) | Q | 80, 240 |
| octanal $C_7H_{15}CHO$ [124-13-0] NUJGJRNETVAIRJ-UHFFFAOYSA-N | $2.0\times10^{-2}$ | 7300 | Brockbank (2013) | L | 1 |
| | $2.1\times10^{-2}$ | | van Ruth et al. (2002) | M | 14 |
| | $1.9\times10^{-2}$ | | van Ruth and Villeneuve (2002) | M | 14, 361 |
| | $8.8\times10^{-3}$ | | van Ruth et al. (2001) | M | 14 |
| | $2.1\times10^{-2}$ | | Li and Carr (1993) | M | |
| | $2.1\times10^{-2}$ | 7400 | Zhou and Mopper (1990) | M | 456 |
| | $1.9\times10^{-2}$ | | Buttery et al. (1969) | M | |
| | $7.5\times10^{-2}$ | | Buttery et al. (1965) | M | |
| | $2.9\times10^{-2}$ | | Amoore and Buttery (1978) | V | |
| | $1.3\times10^{-2}$ | | Yaws (2003) | X | 258 |
| | $1.3\times10^{-2}$ | | Yaws (2003) | X | 237, 38 |
| | $1.9\times10^{-2}$ | | Sieg et al. (2008) | C | |
| | $1.9\times10^{-2}$ | | Nahon et al. (2000) | C | 14 |
| | $2.8\times10^{-2}$ | | Dupeux et al. (2022) | Q | 259 |
| | $7.1\times10^{-2}$ | | Keshavarz et al. (2022) | Q | |





Table A3.5: Aldehydes (RCHO) (. . . continued)

| Substance<br>Formula<br>(Trivial Name)<br>[CAS Registry Number]<br>InChIKey | $H_s^{cp}$<br>(at $T^\ominus$)<br>$\left[\dfrac{\mathrm{mol}}{\mathrm{m^3\,Pa}}\right]$ | $\dfrac{\mathrm{d}\ln H_s^{cp}}{\mathrm{d}(1/T)}$<br><br>[K] | Reference | Type | Note |
|---|---|---|---|---|---|
| | $9.7\times10^{-2}$ | | Duchowicz et al. (2020) | Q | |
| | $2.4\times10^{-3}$ | | Gharagheizi et al. (2012) | Q | |
| | $3.1\times10^{-2}$ | | Raventos-Duran et al. (2010) | Q | 242, 243 |
| | $3.1\times10^{-2}$ | | Raventos-Duran et al. (2010) | Q | 244 |
| | $2.5\times10^{-2}$ | | Raventos-Duran et al. (2010) | Q | 245 |
| | $1.5\times10^{-2}$ | | Gharagheizi et al. (2010) | Q | 246 |
| | $3.9\times10^{-2}$ | | Hilal et al. (2008) | Q | |
| | $7.0\times10^{-2}$ | | Modarresi et al. (2007) | Q | 67 |
| | | 7300 | Kühne et al. (2005) | Q | |
| | $5.2\times10^{-2}$ | | Yaffe et al. (2003) | Q | 248, 272 |
| | $2.0\times10^{-2}$ | | Yao et al. (2002) | Q | 229 |
| | $3.1\times10^{-2}$ | | English and Carroll (2001) | Q | 230, 231 |
| | $4.5\times10^{-2}$ | | Katritzky et al. (1998) | Q | |
| | $3.6\times10^{-2}$ | | Nirmalakhandan et al. (1997) | Q | |
| | $2.8\times10^{-2}$ | | Suzuki et al. (1992) | Q | 232 |
| | $1.9\times10^{-2}$ | | Duchowicz et al. (2020) | ? | 185, 21 |
| | | 6200 | Kühne et al. (2005) | ? | |
| | $1.8\times10^{-2}$ | | Yaws (1999) | ? | 21, 38 |
| | $2.0$ | | Yaws and Yang (1992) | ? | 21, 38 |
| | $1.9\times10^{-2}$ | | Abraham et al. (1990) | ? | |
| 2-ethylhexanal<br>$C_8H_{16}O$<br>[123-05-7]<br>LGYNIFWIKSEESD-UHFFFAOYSA-N | $1.2\times10^{-2}$ | 5400 | Brockbank (2013) | L | 1 |
| | $1.3\times10^{-2}$ | | Duchowicz et al. (2020) | V | 186 |
| | $1.2\times10^{-2}$ | | HSDB (2015) | V | |
| | $2.4\times10^{-2}$ | | Yaws (2003) | X | 258 |
| | $2.4\times10^{-2}$ | | Yaws (2003) | X | 237 |
| | $2.1\times10^{-2}$ | | Dupeux et al. (2022) | Q | 259 |
| | $3.8\times10^{-2}$ | | Duchowicz et al. (2020) | Q | |
| | $3.1\times10^{-2}$ | | Raventos-Duran et al. (2010) | Q | 242, 243 |
| | $2.0\times10^{-2}$ | | Raventos-Duran et al. (2010) | Q | 244 |
| | $2.5\times10^{-2}$ | | Raventos-Duran et al. (2010) | Q | 245 |
| | $2.1\times10^{-2}$ | | Gharagheizi et al. (2010) | Q | 246 |
| | $2.7\times10^{-2}$ | | Hilal et al. (2008) | Q | |
| | $2.4\times10^{-2}$ | | Modarresi et al. (2007) | Q | 67 |
| MCM:C729CHO<br>$C_8H_{12}O_2$<br>MEGRLTONSLAOOM-UHFFFAOYSA-N | $8.0\times10^{2}$ | 10000 | Wieser et al. (2023) | Q | 437 |
| | $7.1\times10^{1}$ | | Wang et al. (2017) | Q | 80, 238 |
| | $1.0\times10^{2}$ | | Wang et al. (2017) | Q | 80, 239 |
| | $6.2\times10^{1}$ | | Wang et al. (2017) | Q | 80, 240 |
| MCM:C810OH<br>$C_8H_{14}O_3$<br>WBGXLUPQADVOIR-UHFFFAOYSA-N | $6.2\times10^{4}$ | | Wang et al. (2017) | Q | 80, 238 |
| | $1.7\times10^{5}$ | | Wang et al. (2017) | Q | 80, 239 |
| | $2.2\times10^{4}$ | | Wang et al. (2017) | Q | 80, 240 |
| MCM:C822OH<br>$C_8H_{14}O_2$<br>CCTLWIOCAVSPLW-UHFFFAOYSA-N | $2.3\times10^{2}$ | | Wang et al. (2017) | Q | 80, 238 |
| | $1.8\times10^{3}$ | | Wang et al. (2017) | Q | 80, 239 |
| | $2.8\times10^{2}$ | | Wang et al. (2017) | Q | 80, 240 |



Table A3.5: Aldehydes (RCHO) (...continued)

| Substance<br>Formula<br>(Trivial Name)<br>[CAS Registry Number]<br>InChIKey | $H_s^{cp}$<br>(at $T^{\ominus}$)<br>$\left[\dfrac{\text{mol}}{\text{m}^3\,\text{Pa}}\right]$ | $\dfrac{\text{d}\ln H_s^{cp}}{\text{d}(1/T)}$<br><br>[K] | Reference | Type | Note |
|---|---|---|---|---|---|
| MCM:C824OH<br>$C_8H_{14}O_3$<br>RBBQERVBTKJTLF-UHFFFAOYSA-N | $3.3\times10^4$<br>$1.9\times10^5$<br>$1.2\times10^3$ | | Wang et al. (2017)<br>Wang et al. (2017)<br>Wang et al. (2017) | Q<br>Q<br>Q | 80, 238<br>80, 239<br>80, 240 |
| MCM:C826OH<br>$C_8H_{14}O_4$<br>LLAUXDLXTPQIPI-UHFFFAOYSA-N | $7.4\times10^7$<br>$3.0\times10^8$<br>$2.6\times10^5$ | | Wang et al. (2017)<br>Wang et al. (2017)<br>Wang et al. (2017) | Q<br>Q<br>Q | 80, 238<br>80, 239<br>80, 240 |
| MCM:C830OH<br>$C_8H_{14}O_2$<br>KODLEZWYEGWLGP-UHFFFAOYSA-N | $2.5\times10^2$<br>$1.6\times10^3$<br>$6.8\times10^1$ | | Wang et al. (2017)<br>Wang et al. (2017)<br>Wang et al. (2017) | Q<br>Q<br>Q | 80, 238<br>80, 239<br>80, 240 |
| MCM:C831OH<br>$C_8H_{14}O_3$<br>YTEDFRLZFCPFBH-UHFFFAOYSA-N | $6.2\times10^4$<br>$2.1\times10^5$<br>$2.3\times10^3$ | | Wang et al. (2017)<br>Wang et al. (2017)<br>Wang et al. (2017) | Q<br>Q<br>Q | 80, 238<br>80, 239<br>80, 240 |
| MCM:C89OH<br>$C_8H_{14}O_2$<br>KMZPPEYMMZMCIQ-UHFFFAOYSA-N | $2.5\times10^2$<br>$2.3\times10^3$<br>$2.2\times10^3$ | | Wang et al. (2017)<br>Wang et al. (2017)<br>Wang et al. (2017) | Q<br>Q<br>Q | 80, 238<br>80, 239<br>80, 240 |
| nonanal<br>$C_8H_{17}CHO$<br>[124-19-6]<br>GYHFUZHODSMOHU-UHFFFAOYSA-N | $1.1\times10^{-2}$<br>$1.0\times10^{-2}$<br>$1.3\times10^{-2}$<br>$7.1\times10^{-2}$<br>$1.3\times10^{-2}$<br>$1.0\times10^{-2}$<br>$1.0\times10^{-2}$<br>$1.4\times10^{-2}$<br>$1.8\times10^{-2}$<br>$9.6\times10^{-2}$<br>$9.7\times10^{-2}$<br>$1.8\times10^{-3}$<br>$2.0\times10^{-2}$<br>$3.9\times10^{-2}$<br>$2.0\times10^{-2}$<br>$1.1\times10^{-2}$<br>$2.4\times10^{-2}$<br>$5.9\times10^{-2}$<br>$1.3\times10^{-2}$<br>$1.8\times10^{-2}$<br>$2.3\times10^{-2}$<br>$2.8\times10^{-2}$<br>$2.1\times10^{-2}$<br>$2.0\times10^{-2}$<br>$1.3\times10^{-2}$<br>$1.5\times10^{-2}$<br>$6.9\times10^{-3}$<br>$1.3\times10^{-2}$ | 6800<br>6700 | Brockbank (2013)<br>Zhou and Mopper (1990)<br>Buttery et al. (1969)<br>Buttery et al. (1965)<br>Amoore and Buttery (1978)<br>Yaws (2003)<br>Yaws (2003)<br>Sieg et al. (2008)<br>Dupeux et al. (2022)<br>Keshavarz et al. (2022)<br>Duchowicz et al. (2020)<br>Gharagheizi et al. (2012)<br>Raventos-Duran et al. (2010)<br>Raventos-Duran et al. (2010)<br>Raventos-Duran et al. (2010)<br>Gharagheizi et al. (2010)<br>Hilal et al. (2008)<br>Modarresi et al. (2007)<br>Yaffe et al. (2003)<br>Yao et al. (2002)<br>English and Carroll (2001)<br>Nirmalakhandan et al. (1997)<br>Suzuki et al. (1992)<br>Meylan and Howard (1991)<br>Duchowicz et al. (2020)<br>Yaws (1999)<br>Yaws and Yang (1992)<br>Abraham et al. (1990) | L<br>M<br>M<br>M<br>V<br>X<br>X<br>C<br>Q<br>Q<br>Q<br>Q<br>Q<br>Q<br>Q<br>Q<br>Q<br>Q<br>Q<br>Q<br>Q<br>Q<br>Q<br>Q<br>?<br>?<br>?<br>? | 1<br>456<br><br><br><br>258<br>237, 38<br><br>259<br><br>184<br><br>242, 243<br>244<br>245<br>246<br><br>67<br>248, 249<br>229<br>230, 231<br><br>232<br><br>185, 21<br>21, 38<br>21, 38<br> |





Table A3.5: Aldehydes (RCHO) (... continued)

| Substance<br>Formula<br>(Trivial Name)<br>[CAS Registry Number]<br>InChIKey | $H_s^{cp}$<br>(at $T^{\ominus}$)<br>$\left[\dfrac{\text{mol}}{\text{m}^3\,\text{Pa}}\right]$ | $\dfrac{\text{d}\ln H_s^{cp}}{\text{d}(1/T)}$<br><br>[K] | Reference | Type | Note |
|---|---|---|---|---|---|
| 3,5,5-trimethylhexanal<br>C$_9$H$_{18}$O<br>[5435-64-3]<br>WTPYRCJDOZVZON-UHFFFAOYSA-N | $2.0\times10^{-2}$ | | HSDB (2015) | Q | 99 |
| decanal<br>C$_9$H$_{19}$CHO<br>[112-31-2]<br>KSMVZQYAVGTKIV-UHFFFAOYSA-N | $6.3\times10^{-3}$ | 8900 | Brockbank (2013) | L | 1 |
| | $4.3\times10^{-3}$ | | Helburn et al. (2008) | M | |
| | $6.0\times10^{-3}$ | 8700 | Zhou and Mopper (1990) | M | 456 |
| | $1.7\times10^{-1}$ | | Buttery et al. (1965) | M | |
| | $1.3\times10^{-2}$ | | Yaws (2003) | X | 258 |
| | $1.3\times10^{-2}$ | | Yaws (2003) | X | 237 |
| | $5.5\times10^{-3}$ | | Sieg et al. (2008) | C | |
| | $1.9\times10^{-2}$ | | Dupeux et al. (2022) | Q | 259 |
| | $1.8\times10^{-1}$ | | Keshavarz et al. (2022) | Q | |
| | $9.7\times10^{-2}$ | | Duchowicz et al. (2020) | Q | 299 |
| | $2.0\times10^{-3}$ | | Gharagheizi et al. (2012) | Q | |
| | $1.6\times10^{-2}$ | | Raventos-Duran et al. (2010) | Q | 271, 243 |
| | $9.9\times10^{-2}$ | | Raventos-Duran et al. (2010) | Q | 244 |
| | $1.6\times10^{-2}$ | | Raventos-Duran et al. (2010) | Q | 245 |
| | $8.5\times10^{-3}$ | | Gharagheizi et al. (2010) | Q | 246 |
| | $2.6\times10^{-2}$ | | Hilal et al. (2008) | Q | |
| | $5.1\times10^{-2}$ | | Modarresi et al. (2007) | Q | 67 |
| | | 7900 | Kühne et al. (2005) | Q | |
| | $5.6\times10^{-3}$ | | Yaffe et al. (2003) | Q | 248, 249 |
| | $3.8\times10^{-2}$ | | Katritzky et al. (1998) | Q | |
| | $5.5\times10^{-3}$ | | Duchowicz et al. (2020) | ? | 185, 21 |
| | | 8500 | Kühne et al. (2005) | ? | |
| MCM:C918CHO<br>C$_{10}$H$_{16}$O$_2$<br>GWMMOZICECYZLL-UHFFFAOYSA-N | $1.8\times10^{1}$ | | Wang et al. (2017) | Q | 80, 238 |
| | $1.1\times10^{2}$ | | Wang et al. (2017) | Q | 80, 239 |
| | $1.5\times10^{1}$ | | Wang et al. (2017) | Q | 80, 240 |
| hydroxycitronellal<br>C$_{10}$H$_{20}$O$_2$<br>[107-75-5]<br>WPFVBOQKRVRMJB-UHFFFAOYSA-N | $9.4\times10^{2}$ | | Dupeux et al. (2022) | Q | 259 |
| undecanal<br>C$_{11}$H$_{22}$O<br>[112-44-7]<br>KMPQYAYAQWNLME-UHFFFAOYSA-N | $7.0\times10^{-3}$ | | Yaws (2003) | X | 258 |
| | $6.9\times10^{-3}$ | | Yaws (2003) | X | 237 |
| | $1.5\times10^{-2}$ | | Dupeux et al. (2022) | Q | 259 |
| | $2.2\times10^{-3}$ | | Gharagheizi et al. (2012) | Q | |
| | $6.9\times10^{-3}$ | | Gharagheizi et al. (2010) | Q | 246 |
| | | 8300 | Kühne et al. (2005) | Q | |
| | | 8300 | Kühne et al. (2005) | ? | |





Table A3.5: Aldehydes (RCHO) (. . . continued)

| Substance<br>Formula<br>(Trivial Name)<br>[CAS Registry Number]<br>InChIKey | $H_s^{cp}$<br>(at $T^{\ominus}$)<br>$\left[\dfrac{\mathrm{mol}}{\mathrm{m}^3\,\mathrm{Pa}}\right]$ | $\dfrac{\mathrm{d}\ln H_s^{cp}}{\mathrm{d}(1/T)}$<br><br>[K] | Reference | Type | Note |
|---|---|---|---|---|---|
| dodecanal<br>$C_{12}H_{24}O$<br>[112-54-9]<br>HFJRKMMYBMWEAD-UHFFFAOYSA-N | $5.2\times10^{-3}$<br>$5.2\times10^{-3}$<br>$1.2\times10^{-2}$<br>$2.3\times10^{-3}$<br>$6.1\times10^{-3}$ | | Yaws (2003)<br>Yaws (2003)<br>Dupeux et al. (2022)<br>Gharagheizi et al. (2012)<br>Gharagheizi et al. (2010) | X<br>X<br>Q<br>Q<br>Q | 258<br>237<br>259<br><br>246 |
| 1-tridecanal<br>$C_{13}H_{26}O$<br>[10486-19-8]<br>BGEHHAVMRVXCGR-UHFFFAOYSA-N | $5.8\times10^{-3}$<br>$5.8\times10^{-3}$<br>$9.6\times10^{-3}$<br>$2.4\times10^{-3}$<br>$5.9\times10^{-3}$ | | Yaws (2003)<br>Yaws (2003)<br>Dupeux et al. (2022)<br>Gharagheizi et al. (2012)<br>Gharagheizi et al. (2010) | X<br>X<br>Q<br>Q<br>Q | 258<br>237<br>259<br><br>246 |
| 1-tetradecanal<br>$C_{14}H_{28}O$<br>[124-25-4]<br>UHUFTBALEZWWIH-UHFFFAOYSA-N | $8.0\times10^{-4}$<br>$6.2\times10^{-3}$<br>$2.6\times10^{-3}$ | | Yaws (2003)<br>Dupeux et al. (2022)<br>Gharagheizi et al. (2012) | X<br>Q<br>Q | 258<br>259<br> |
| aquaflora<br>$C_{12}H_{18}O$<br>[1339119-15-1]<br>BFTAXJRKNXWMMX-UHFFFAOYSA-N | $5.2\times10^{-1}$ | | Dupeux et al. (2022) | Q | 259 |
| starfleur<br>$C_{13}H_{24}O$<br>(3-(4-isobutylcyclohexyl)propanal)<br>[1254940-85-6]<br>XUTZHJNXAOHUTM-UHFFFAOYSA-N | $1.4\times10^{-1}$ | | Dupeux et al. (2022) | Q | 259 |
| MCM:C126CO<br>$C_{12}H_{18}O_2$<br>IWMJDJFLMSAAES-UHFFFAOYSA-N | $8.5\times10^{1}$<br>$1.7\times10^{2}$<br>$1.7\times10^{2}$ | | Wang et al. (2017)<br>Wang et al. (2017)<br>Wang et al. (2017) | Q<br>Q<br>Q | 80, 238<br>80, 239<br>80, 240 |
| MCM:C126OH<br>$C_{12}H_{20}O_2$<br>PQYIIYBUHROKIY-UHFFFAOYSA-N | $2.7\times10^{2}$<br>$2.1\times10^{3}$<br>$4.4\times10^{2}$ | | Wang et al. (2017)<br>Wang et al. (2017)<br>Wang et al. (2017) | Q<br>Q<br>Q | 80, 238<br>80, 239<br>80, 240 |
| MCM:C127OH<br>$C_{12}H_{20}O_3$<br>QHKXYVNBHFONML-UHFFFAOYSA-N | $5.0\times10^{5}$<br>$7.1\times10^{6}$<br>$3.4\times10^{2}$ | | Wang et al. (2017)<br>Wang et al. (2017)<br>Wang et al. (2017) | Q<br>Q<br>Q | 80, 238<br>80, 239<br>80, 240 |
| MCM:C126CHO<br>$C_{13}H_{20}O_2$<br>ZDIXDRBTLRZYNM-UHFFFAOYSA-N | $6.9\times10^{1}$<br>$1.1\times10^{2}$<br>$4.9\times10^{2}$ | | Wang et al. (2017)<br>Wang et al. (2017)<br>Wang et al. (2017) | Q<br>Q<br>Q | 80, 238<br>80, 239<br>80, 240 |
| MCM:C1311OH<br>$C_{13}H_{22}O_3$<br>CBUCUSJJCJUTML-UHFFFAOYSA-N | $4.0\times10^{5}$<br>$8.3\times10^{6}$<br>$6.5\times10^{3}$ | | Wang et al. (2017)<br>Wang et al. (2017)<br>Wang et al. (2017) | Q<br>Q<br>Q | 80, 238<br>80, 239<br>80, 240 |
| MCM:C1313OH<br>$C_{13}H_{22}O_4$<br>XIWFBNXBOAYHAJ-UHFFFAOYSA-N | $7.3\times10^{7}$<br>$4.3\times10^{8}$<br>$8.3\times10^{4}$ | | Wang et al. (2017)<br>Wang et al. (2017)<br>Wang et al. (2017) | Q<br>Q<br>Q | 80, 238<br>80, 239<br>80, 240 |





Table A3.5: Aldehydes (RCHO) (... continued)

| Substance Formula (Trivial Name) [CAS Registry Number] InChIKey | $H_s^{cp}$ (at $T^\ominus$) $\left[\dfrac{\mathrm{mol}}{\mathrm{m^3\,Pa}}\right]$ | $\dfrac{\mathrm{d}\ln H_s^{cp}}{\mathrm{d}(1/T)}$ [K] | Reference | Type | Note |
|---|---|---|---|---|---|
| MCM:C136OH | $2.1\times10^{2}$ | | Wang et al. (2017) | Q | 80, 238 |
| $C_{13}H_{22}O_2$ | $2.0\times10^{3}$ | | Wang et al. (2017) | Q | 80, 239 |
| GEHGNVSPJQGFQG-UHFFFAOYSA-N | $1.0\times10^{3}$ | | Wang et al. (2017) | Q | 80, 240 |
| propenal | $1.3\times10^{-1}$ | 5500 | Burkholder et al. (2019) | L | 469 |
| CH$_2$CHCHO | $7.2\times10^{-2}$ | 5100 | Brockbank (2013) | L | 1 |
| (acrolein) | $7.2\times10^{-2}$ | 5100 | Snider and Dawson (1985) | M | |
| [107-02-8] | $1.0\times10^{-1}$ | | Mackay et al. (2006c) | V | |
| HGINCPLSRVDWNT-UHFFFAOYSA-N | 2.3 | | Lide and Frederikse (1995) | V | |
| | $1.0\times10^{-2}$ | | Mackay et al. (1995) | V | |
| | $7.0\times10^{-2}$ | | Hwang et al. (1992) | V | |
| | $1.3\times10^{-1}$ | | Suntio et al. (1988) | V | 12 |
| | $1.0\times10^{-1}$ | 3800 | Goldstein (1982) | X | 298 |
| | 2.2 | | Howard (1989) | X | 412 |
| | $8.1\times10^{-2}$ | | Gaffney and Senum (1984) | X | 389 |
| | $1.8\times10^{-1}$ | | Suntio et al. (1988) | C | 12 |
| | $1.4\times10^{-1}$ | | Ryan et al. (1988) | C | |
| | $1.2\times10^{-1}$ | | Keshavarz et al. (2022) | Q | |
| | $3.5\times10^{-1}$ | | Duchowicz et al. (2020) | Q | |
| | $2.2\times10^{-1}$ | | Wang et al. (2017) | Q | 80, 238 |
| | $1.9\times10^{-1}$ | | Wang et al. (2017) | Q | 80, 239 |
| | $2.1\times10^{-1}$ | | Wang et al. (2017) | Q | 80, 240 |
| | $5.2\times10^{-2}$ | | Gharagheizi et al. (2012) | Q | |
| | $6.2\times10^{-2}$ | | Raventos-Duran et al. (2010) | Q | 242, 243 |
| | $1.2\times10^{-1}$ | | Raventos-Duran et al. (2010) | Q | 244 |
| | $3.1\times10^{-1}$ | | Raventos-Duran et al. (2010) | Q | 245 |
| | $9.5\times10^{-2}$ | | Hilal et al. (2008) | Q | |
| | $6.3\times10^{-1}$ | | Modarresi et al. (2007) | Q | 67 |
| | | 4600 | Kühne et al. (2005) | Q | |
| | $8.1\times10^{-2}$ | | Duchowicz et al. (2020) | ? | 185, 21 |
| | $7.5\times10^{-2}$ | | Mackay et al. (2006c) | ? | 21 |
| | | 3800 | Kühne et al. (2005) | ? | |
| | $1.2\times10^{-1}$ | | Yaws (1999) | ? | 21, 12 |
| 2-methylpropenal | $4.5\times10^{-2}$ | 4600 | Burkholder et al. (2019) | L | |
| $C_4H_6O$ | $6.4\times10^{-2}$ | | Burkholder et al. (2015) | L | |
| (methacrolein) | $4.5\times10^{-2}$ | 4600 | Brockbank (2013) | L | 1 |
| [78-85-3] | $4.8\times10^{-2}$ | 4300 | Ji and Evans (2007) | M | |
| STNJBCKSHOAVAJ-UHFFFAOYSA-N | $6.4\times10^{-2}$ | | Iraci et al. (1999) | M | |
| | $4.2\times10^{-2}$ | 5300 | Allen et al. (1998) | M | |
| | $5.2\times10^{-2}$ | | HSDB (2015) | V | |
| | $4.7\times10^{-2}$ | | Yaws (2003) | X | 258 |
| | $4.7\times10^{-2}$ | | Yaws (2003) | X | 237 |
| | $7.3\times10^{-2}$ | | Dupeux et al. (2022) | Q | 259 |
| | $1.6\times10^{-1}$ | | Keshavarz et al. (2022) | Q | |
| | $1.1\times10^{-1}$ | | Duchowicz et al. (2020) | Q | 184 |
| | $1.5\times10^{-1}$ | | Wang et al. (2017) | Q | 80, 238 |
| | $1.4\times10^{-1}$ | | Wang et al. (2017) | Q | 80, 239 |



Table A3.5: Aldehydes (RCHO) (... continued)

| Substance Formula (Trivial Name) [CAS Registry Number] InChIKey | $H_s^{cp}$ (at $T^{\ominus}$) $\left[\dfrac{\mathrm{mol}}{\mathrm{m^3\,Pa}}\right]$ | $\dfrac{\mathrm{d}\ln H_s^{cp}}{\mathrm{d}(1/T)}$ [K] | Reference | Type | Note |
|---|---|---|---|---|---|
| | $8.3\times10^{-2}$ | | Wang et al. (2017) | Q | 80, 240 |
| | $1.6\times10^{-2}$ | | Gharagheizi et al. (2012) | Q | |
| | $3.9\times10^{-2}$ | | Raventos-Duran et al. (2010) | Q | 242, 243 |
| | $9.9\times10^{-2}$ | | Raventos-Duran et al. (2010) | Q | 244 |
| | $1.6\times10^{-1}$ | | Raventos-Duran et al. (2010) | Q | 245 |
| | $4.7\times10^{-2}$ | | Gharagheizi et al. (2010) | Q | 246 |
| | $9.5\times10^{-2}$ | | Hilal et al. (2008) | Q | |
| | $1.1\times10^{-1}$ | | Modarresi et al. (2007) | Q | 67 |
| | | 4000 | Kühne et al. (2005) | Q | |
| | $4.3\times10^{-2}$ | | Duchowicz et al. (2020) | ? | 185, 21 |
| | | 4800 | Kühne et al. (2005) | ? | |
| | $4.8\times10^{-2}$ | | Yaws (1999) | ? | 21 |
| 2-butenal $C_4H_6O$ [4170-30-3] MLUCVPSAIODCQM-UHFFFAOYSA-N | $2.9\times10^{-1}$ | | Wang et al. (2017) | Q | 80, 238 |
| | $1.6\times10^{-1}$ | | Wang et al. (2017) | Q | 80, 239 |
| | 1.6 | | Wang et al. (2017) | Q | 80, 240 |
| | $9.7\times10^{-2}$ | | Hilal et al. (2008) | Q | |
| | $3.9\times10^{-1}$ | | Modarresi et al. (2007) | Q | 67 |
| | $2.7\times10^{-1}$ | | Nirmalakhandan et al. (1997) | Q | |
| | | | Burkholder et al. (2019) | W | 470 |
| $(E)$-2-butenal $CH_3CHCHCHO$ (crotonaldehyde) [123-73-9] MLUCVPSAIODCQM-NSCUHMNNSA-N | $5.4\times10^{-1}$ | 5300 | Brockbank (2013) | L | 1 |
| | $5.0\times10^{-1}$ | | Buttery et al. (1971) | M | |
| | $4.4\times10^{-2}$ | | Mackay et al. (2006c) | V | |
| | $4.4\times10^{-2}$ | | Mackay et al. (1995) | V | |
| | $7.6\times10^{-1}$ | | Yaws (2003) | X | 258 |
| | $5.9\times10^{-1}$ | 3600 | Goldstein (1982) | X | 298 |
| | $5.0\times10^{-1}$ | | Gaffney and Senum (1984) | X | 389 |
| | 1.5 | | Dupeux et al. (2022) | Q | 259 |
| | $1.6\times10^{-1}$ | | Keshavarz et al. (2022) | Q | |
| | $1.2\times10^{-1}$ | | Duchowicz et al. (2020) | Q | 299 |
| | $3.9\times10^{-1}$ | | Modarresi et al. (2007) | Q | 67 |
| | | 5000 | Kühne et al. (2005) | Q | |
| | $5.2\times10^{-1}$ | | Yaffe et al. (2003) | Q | 248, 249 |
| | $3.1\times10^{-1}$ | | Suzuki et al. (1992) | Q | 232 |
| | $5.1\times10^{-1}$ | | Duchowicz et al. (2020) | ? | 185, 21 |
| | | 4300 | Kühne et al. (2005) | ? | |
| | $5.9\times10^{-1}$ | | Yaws (1999) | ? | 21, 12 |
| | $5.1\times10^{-1}$ | | Abraham et al. (1990) | ? | |
| 2-hexenal $C_6H_{10}O$ [505-57-7] MBDOYVRWFFCFHM-UHFFFAOYSA-N | $3.0\times10^{-1}$ | | Keshavarz et al. (2022) | Q | |
| | $1.2\times10^{-1}$ | | Duchowicz et al. (2020) | Q | |
| | $6.2\times10^{-2}$ | | Hilal et al. (2008) | Q | |
| | $2.5\times10^{-1}$ | | Modarresi et al. (2007) | Q | 67 |
| | $8.6\times10^{-2}$ | | English and Carroll (2001) | Q | 230, 260 |
| | $1.7\times10^{-1}$ | | Nirmalakhandan et al. (1997) | Q | |
| | $2.0\times10^{-1}$ | | Duchowicz et al. (2020) | ? | 185, 21 |





Table A3.5: Aldehydes (RCHO) (. . . continued)

| Substance<br>Formula<br>(Trivial Name)<br>[CAS Registry Number]<br>InChIKey | $H_s^{cp}$<br>(at $T^\ominus$)<br>$\left[\dfrac{\mathrm{mol}}{\mathrm{m^3\,Pa}}\right]$ | $\dfrac{\mathrm{d}\ln H_s^{cp}}{\mathrm{d}(1/T)}$<br><br>[K] | Reference | Type | Note |
|---|---|---|---|---|---|
| (E)-2-hexenal<br>C$_3$H$_7$CHCHCHO<br>(trans-2-hexenal)<br>[6728-26-3]<br>MBDOYVRWFFCFHM-SNAWJCMRSA-N | $1.4\times10^{-1}$<br>$9.0\times10^{-1}$<br>$2.0\times10^{-1}$<br>$2.0\times10^{-1}$<br>$2.1\times10^{-1}$<br>$1.8\times10^{-1}$ | 5700 | Karl et al. (2003)<br>Meynier et al. (2003)<br>Buttery et al. (1971)<br>Meynier et al. (2003)<br>Yaffe et al. (2003)<br>Suzuki et al. (1992) | M<br>M<br>M<br>C<br>Q<br>Q | <br>38<br><br><br>248, 249<br>232 |
| (E,E)-2,4-hexadienal<br>CH$_3$CHCHCHCHCHO<br>(trans-trans-2,4-hexadienal)<br>[142-83-6]<br>BATOPAZDIZEVQF-MQQKCMAXSA-N | 1.0<br>$3.0\times10^{-1}$<br>$1.4\times10^{-1}$<br>$3.9\times10^{-1}$<br>$4.7\times10^{-1}$<br>1.5<br>1.0 | | Buttery et al. (1971)<br>Keshavarz et al. (2022)<br>Duchowicz et al. (2020)<br>Hilal et al. (2008)<br>Modarresi et al. (2007)<br>Suzuki et al. (1992)<br>Duchowicz et al. (2020) | M<br>Q<br>Q<br>Q<br>Q<br>Q<br>? | <br><br>184<br><br>67<br>232<br>185, 21 |
| 2-heptenal<br>C$_7$H$_{12}$O<br>[2463-63-0]<br>NDFKTBCGKNOHPJ-UHFFFAOYSA-N | $5.0\times10^{-2}$ | | Hilal et al. (2008) | Q | |
| trans-2-heptenal<br>C$_7$H$_{12}$O<br>[18829-55-5]<br>NDFKTBCGKNOHPJ-AATRIKPKSA-N | 1.3 | | Abney (2021) | Q | 399 |
| (Z)-4-heptenal<br>C$_7$H$_{12}$O<br>(cis-4-heptenal)<br>[6728-31-0]<br>VVGOCOMZRGWHPI-ARJAWSKDSA-N | $8.8\times10^{-2}$ | | Straver and de Loos (2005) | M | |
| 2-octenal<br>C$_8$H$_{14}$O<br>[2363-89-5]<br>LVBXEMGDVWVTGY-UHFFFAOYSA-N | $7.1\times10^{-2}$<br>$1.2\times10^{-1}$<br>$4.1\times10^{-2}$<br>$1.9\times10^{-1}$<br>$5.2\times10^{-2}$<br>$1.0\times10^{-1}$<br>$1.3\times10^{-1}$ | | Keshavarz et al. (2022)<br>Duchowicz et al. (2020)<br>Hilal et al. (2008)<br>Modarresi et al. (2007)<br>English and Carroll (2001)<br>Nirmalakhandan et al. (1997)<br>Duchowicz et al. (2020) | Q<br>Q<br>Q<br>Q<br>Q<br>Q<br>? | <br>299<br><br>67<br>230, 231<br><br>185, 21 |
| (E)-2-octenal<br>C$_8$H$_{14}$O<br>(trans-2-octenal)<br>[2548-87-0]<br>LVBXEMGDVWVTGY-VOTSOKGWSA-N | $1.3\times10^{-1}$<br>$1.4\times10^{-1}$<br>$1.1\times10^{-1}$ | | Buttery et al. (1971)<br>Yaffe et al. (2003)<br>Suzuki et al. (1992)<br>Betterton (1992) | M<br>Q<br>Q<br>W | <br>248, 249<br>232<br>471 |
| 2-ethyl-2-hexenal<br>C$_8$H$_{14}$O<br>[645-62-5]<br>PYLMCYQHBRSDND-UHFFFAOYSA-N | $5.2\times10^{-2}$ | 5800 | Brockbank (2013) | L | 1 |



Table A3.5: Aldehydes (RCHO) (... continued)

| Substance Formula (Trivial Name) [CAS Registry Number] InChIKey | $H_s^{cp}$ (at $T^{\ominus}$) $\left[\dfrac{\text{mol}}{\text{m}^3\,\text{Pa}}\right]$ | $\dfrac{\text{d}\ln H_s^{cp}}{\text{d}(1/T)}$ [K] | Reference | Type | Note |
|---|---|---|---|---|---|
| (E)-2-nonenal $C_9H_{16}O$ (trans-2-nonenal) [18829-56-6] BSAIUMLZVGUGKX-BQYQJAHWSA-N | $5.8\times10^{-2}$ | | Roberts and Pollien (1997) | M | |
| trans-2,cis-6-nonadienal $C_9H_{14}O$ [557-48-2] HZYHMHHBBBSGHB-ODYTWBPASA-N | $8.2\times10^{-2}$ | 7100 | Ömür-Özbek and Dietrich (2005) | M | |
| 3,7-dimethyl-6-octenal $C_{10}H_{18}O$ (citronellal) [106-23-0] NEHNMFOYXAPHSD-UHFFFAOYSA-N | $2.5\times10^{-2}$ $3.8\times10^{-2}$ | 4500 | van Roon et al. (2005) HSDB (2015) | V Q | 99 |
| 3,7-dimethyl-2,6-octadienal $C_{10}H_{16}O$ (citral) [5392-40-5] WTEVQBCEXWBHNA-UHFFFAOYSA-N | $2.9\times10^{-1}$ $5.2$ $2.3\times10^{-1}$ | 6700 | Wu et al. (2022b) Dupeux et al. (2022) HSDB (2015) | M Q Q | 259 99 |
| tillenal $C_{11}H_{18}O$ (3-(4,4-dimethyl-1-cyclohexen-1-yl)propanal) [850997-10-3] IHMKWBJKOWYASH-UHFFFAOYSA-N | $3.4\times10^{-1}$ | | Dupeux et al. (2022) | Q | 259 |
| lilybelle $C_{12}H_{20}O$ [1378867-81-2] VZZSYXAVGYODQG-UHFFFAOYSA-N | $4.5\times10^{-1}$ | | Dupeux et al. (2022) | Q | 259 |
| mugoxal $C_{13}H_{22}O$ (3-(4-tert-butylcyclohexen-1-yl)propanal) WGWWNPFQPHLSIM-UHFFFAOYSA-N | $6.7\times10^{-1}$ | | Dupeux et al. (2022) | Q | 259 |
| orange oil $C_{15}H_{22}O$ [8028-48-6] NOPLRNXKHZRXHT-UHFFFAOYSA-N | $6.3\times10^{-4}$ | | Maniere et al. (2011) | ? | 241, 165 |



Table A3.5: Aldehydes (RCHO) (. . . continued)

| Substance Formula (Trivial Name) [CAS Registry Number] InChIKey | $H_s^{cp}$ (at $T^\ominus$) $\left[\dfrac{\text{mol}}{\text{m}^3\,\text{Pa}}\right]$ | $\dfrac{\text{d}\ln H_s^{cp}}{\text{d}(1/T)}$ [K] | Reference | Type | Note |
|---|---|---|---|---|---|
| benzaldehyde | $4.0\times10^{-1}$ | 5200 | Brockbank (2013) | L | 1, 472 |
| $C_6H_5CHO$ | $3.8\times10^{-1}$ | 5500 | Staudinger and Roberts (2001) | L | |
| [100-52-7] | $3.9\times10^{-1}$ | 4800 | Staudinger and Roberts (1996) | L | |
| HUMNYLRZRPPJDN-UHFFFAOYSA-N | $3.2\times10^{-1}$ | 6300 | Allou et al. (2011) | M | |
| | $3.4\times10^{-1}$ | | Souchon et al. (2004) | M | |
| | $3.5\times10^{-1}$ | 7000 | Allen et al. (1998) | M | |
| | $4.2\times10^{-1}$ | 4600 | Zhou and Mopper (1990) | M | 456 |
| | $3.7\times10^{-1}$ | 5100 | Betterton and Hoffmann (1988) | M | 460 |
| | $1.6\times10^{-1}$ | | Mackay et al. (2006c) | V | |
| | $1.6\times10^{-1}$ | | Mackay et al. (1995) | V | |
| | $3.6\times10^{-1}$ | | Hine and Mookerjee (1975) | V | |
| | $3.5\times10^{-1}$ | 5400 | Bagno et al. (1991) | T | 473 |
| | $3.9\times10^{-1}$ | | Yaws (2003) | X | 258 |
| | $3.6\times10^{-1}$ | | Gaffney and Senum (1984) | X | 389 |
| | $3.7\times10^{-1}$ | | Schüürmann (2000) | C | 21 |
| | $8.4\times10^{-1}$ | | Dupeux et al. (2022) | Q | 259 |
| | $5.3\times10^{-2}$ | | Keshavarz et al. (2022) | Q | |
| | $6.6\times10^{-1}$ | | Duchowicz et al. (2020) | Q | 184 |
| | 3.0 | | Wang et al. (2017) | Q | 80, 238 |
| | 1.3 | | Wang et al. (2017) | Q | 80, 239 |
| | 1.6 | | Wang et al. (2017) | Q | 80, 240 |
| | $3.6\times10^{-1}$ | | Li et al. (2014) | Q | 241 |
| | $3.9\times10^{-1}$ | | Raventos-Duran et al. (2010) | Q | 271, 243 |
| | $4.9\times10^{-1}$ | | Raventos-Duran et al. (2010) | Q | 244 |
| | $7.8\times10^{-1}$ | | Raventos-Duran et al. (2010) | Q | 245 |
| | $7.7\times10^{-1}$ | | Hilal et al. (2008) | Q | |
| | 1.2 | | Modarresi et al. (2007) | Q | 67 |
| | $2.6\times10^{-2}$ | | Emel'yanenko et al. (2007) | Q | 415 |
| | $2.6\times10^{-2}$ | | Hertel and Sommer (2006) | Q | 415 |
| | | 5800 | Kühne et al. (2005) | Q | |
| | $3.7\times10^{-1}$ | | Yaffe et al. (2003) | Q | 248, 249 |
| | $5.4\times10^{-1}$ | | English and Carroll (2001) | Q | 230, 274 |
| | $2.4\times10^{-1}$ | | Katritzky et al. (1998) | Q | |
| | $7.2\times10^{-1}$ | | Nirmalakhandan et al. (1997) | Q | |
| | $3.6\times10^{-1}$ | | Suzuki et al. (1992) | Q | 232 |
| | $3.7\times10^{-1}$ | | Duchowicz et al. (2020) | ? | 185, 21 |
| | $4.4\times10^{-1}$ | | Mackay et al. (2006c) | ? | 21 |
| | | 5400 | Kühne et al. (2005) | ? | |
| | $4.0\times10^{-1}$ | | Yaws (1999) | ? | 21 |
| | $3.6\times10^{-1}$ | | Abraham et al. (1990) | ? | |
| phenylacetaldehyde | 1.6 | | Dupeux et al. (2022) | Q | 259 |
| $C_6H_5CH_2CHO$ | 2.8 | | Wang et al. (2017) | Q | 80, 238 |
| [122-78-1] | 3.6 | | Wang et al. (2017) | Q | 80, 239 |
| DTUQWGWMVIHBKE-UHFFFAOYSA-N | 3.6 | | Wang et al. (2017) | Q | 80, 240 |
| | $1.0\times10^{-1}$ | | Emel'yanenko et al. (2007) | Q | 415 |
| | $1.0\times10^{-1}$ | | Hertel and Sommer (2005) | Q | 415 |



Table A3.5: Aldehydes (RCHO) (. . . continued)

| Substance Formula (Trivial Name) [CAS Registry Number] InChIKey | $H_s^{cp}$ (at $T^\ominus$) $\left[\dfrac{\mathrm{mol}}{\mathrm{m^3\,Pa}}\right]$ | $\dfrac{\mathrm{d\ln} H_s^{cp}}{\mathrm{d}(1/T)}$ [K] | Reference | Type | Note |
|---|---|---|---|---|---|
| 2-methylbenzaldehyde | $3.1\times10^{-1}$ | 6900 | Ji et al. (2008) | M | |
| $C_8H_8O$ | 1.9 | | Wang et al. (2017) | Q | 80, 238 |
| ($o$-tolualdehyde) | 1.7 | | Wang et al. (2017) | Q | 80, 239 |
| [529-20-4] | $8.7\times10^{-1}$ | | Wang et al. (2017) | Q | 80, 240 |
| BTFQKIATRPGRBS-UHFFFAOYSA-N | $3.3\times10^{-1}$ | | HSDB (2015) | Q | 99 |
| 3-methylbenzaldehyde | $3.5\times10^{-1}$ | 7200 | Brockbank (2013) | L | 1 |
| $C_8H_8O$ | $3.0\times10^{-1}$ | 5800 | Wu et al. (2022b) | M | |
| ($m$-tolualdehyde) | $3.5\times10^{-1}$ | 7200 | Ji et al. (2008) | M | |
| [620-23-5] | 1.9 | | Wang et al. (2017) | Q | 80, 238 |
| OVWYEQOVUDKZNU-UHFFFAOYSA-N | 1.1 | | Wang et al. (2017) | Q | 80, 239 |
| | 1.7 | | Wang et al. (2017) | Q | 80, 240 |
| | $3.3\times10^{-1}$ | | HSDB (2015) | Q | 99 |
| 4-methylbenzaldehyde | $5.3\times10^{-1}$ | 7200 | Brockbank (2013) | L | 1 |
| $C_8H_8O$ | $4.6\times10^{-1}$ | 6400 | Wu et al. (2022b) | M | |
| ($p$-tolualdehyde) | $5.3\times10^{-1}$ | 7200 | Ji et al. (2008) | M | |
| [104-87-0] | $5.7\times10^{-1}$ | | Duchowicz et al. (2020) | V | 186 |
| FXLOVSHXALFLKQ-UHFFFAOYSA-N | $5.8\times10^{-1}$ | | HSDB (2015) | V | |
| | $5.4\times10^{-1}$ | | Abraham et al. (1994a) | R | |
| | $3.3\times10^{-1}$ | | Duchowicz et al. (2020) | Q | |
| | 1.9 | | Wang et al. (2017) | Q | 80, 238 |
| | 1.3 | | Wang et al. (2017) | Q | 80, 239 |
| | 3.0 | | Wang et al. (2017) | Q | 80, 240 |
| | $7.9\times10^{-1}$ | | Hilal et al. (2008) | Q | |
| | $5.3\times10^{-1}$ | | Modarresi et al. (2007) | Q | 67 |
| | $4.8\times10^{-1}$ | | English and Carroll (2001) | Q | 230, 231 |
| | $6.5\times10^{-1}$ | | Katritzky et al. (1998) | Q | |
| | $5.2\times10^{-1}$ | | Nirmalakhandan et al. (1997) | Q | |
| | $5.6\times10^{-1}$ | | Yaws (1999) | ? | 21 |
| terephthaldialdehyde $C_8H_6O_2$ [623-27-8] KUCOHFSKRZZVRO-UHFFFAOYSA-N | $1.6\times10^{1}$ | | Abraham et al. (2019) | Q | |
| 2-hydroxybenzaldehyde | 1.1 | 6200 | Ji et al. (2008) | M | |
| $C_6H_4(OH)CHO$ | 1.8 | | Duchowicz et al. (2020) | V | 186 |
| (2-formylphenol; salicylaldehyde) | 4.6 | | Duchowicz et al. (2020) | Q | |
| [90-02-8] | $4.0\times10^{3}$ | | McFall et al. (2020) | Q | 474 |
| SMQUZDBALVYZAC-UHFFFAOYSA-N | 6.2 | | Raventos-Duran et al. (2010) | Q | 271, 243 |
| | 9.9 | | Raventos-Duran et al. (2010) | Q | 244 |
| | 6.2 | | Raventos-Duran et al. (2010) | Q | 245 |
| | $1.6\times10^{1}$ | | Hilal et al. (2008) | Q | |
| | 1.8 | | Yaws (1999) | ? | 21, 475 |



Table A3.5: Aldehydes (RCHO) (... continued)

| Substance Formula (Trivial Name) [CAS Registry Number] InChIKey | $H_s^{cp}$ (at $T^{\ominus}$) $\left[\dfrac{\text{mol}}{\text{m}^3\,\text{Pa}}\right]$ | $\dfrac{\text{d}\ln H_s^{cp}}{\text{d}(1/T)}$ [K] | Reference | Type | Note |
|---|---|---|---|---|---|
| 3-hydroxybenzaldehyde $C_6H_4(OH)CHO$ (3-formylphenol) [100-83-4] IAVREABSGIHHMO-UHFFFAOYSA-N | $3.9\times10^3$ $9.6\times10^3$ $1.8\times10^4$ $1.2\times10^3$ $1.6\times10^4$ $6.2\times10^3$ $5.3\times10^3$ $7.9\times10^2$ $1.2\times10^3$ $3.0\times10^4$ $3.9\times10^3$ $3.8\times10^3$ | | Gaffney and Senum (1984) Keshavarz et al. (2022) Duchowicz et al. (2020) Raventos-Duran et al. (2010) Raventos-Duran et al. (2010) Raventos-Duran et al. (2010) Hilal et al. (2008) Modarresi et al. (2007) Katritzky et al. (1998) Nirmalakhandan et al. (1997) Duchowicz et al. (2020) Abraham et al. (1990) | X Q Q Q Q Q Q Q Q Q ? ? | 389 299 242, 243 244 245 67 185, 21 |
| 4-hydroxybenzaldehyde $C_6H_4(OH)CHO$ (4-formylphenol) [123-08-0] RGHHSNMVTDWUBI-UHFFFAOYSA-N | $1.9\times10^4$ $9.6\times10^3$ $2.1\times10^4$ $1.2\times10^3$ $1.2\times10^3$ $2.5\times10^3$ $6.2\times10^3$ $8.8\times10^2$ $1.6\times10^3$ $3.0\times10^4$ $1.9\times10^4$ $1.9\times10^4$ | 8600 | Parsons et al. (1971) Keshavarz et al. (2022) Duchowicz et al. (2020) Gharagheizi et al. (2012) Raventos-Duran et al. (2010) Raventos-Duran et al. (2010) Raventos-Duran et al. (2010) Hilal et al. (2008) Modarresi et al. (2007) Nirmalakhandan et al. (1997) Duchowicz et al. (2020) Abraham et al. (1990) | T Q Q Q Q Q Q Q Q Q ? ? | 417 242, 243 244 245 67 185, 21 |
| 2-methoxybenzaldehyde $C_8H_8O_2$ [135-02-4] PKZJLOCLABXVMC-UHFFFAOYSA-N | $1.0\times10^1$ | 8900 | Ji et al. (2008) | M | |
| 3-methoxybenzaldehyde $C_8H_8O_2$ [591-31-1] WMPDAIZRQDCGFH-UHFFFAOYSA-N | 5.1 | 8800 | Ji et al. (2008) | M | |
| 4-methoxybenzaldehyde $C_8H_8O_2$ [123-11-5] ZRSNZINYAWTAHE-UHFFFAOYSA-N | $2.4\times10^1$ | 8700 | Ji et al. (2008) | M | |
| 3-phenyl-2-propenal $C_9H_8O$ (cinnamaldehyde) [104-55-2] KJPRLNWUNMBNBZ-QPJJXVBHSA-N | 2.8 1.4 | 6300 | HSDB (2015) van Roon et al. (2005) | V V | |





Table A3.5: Aldehydes (RCHO) (...continued)

| Substance<br>Formula<br>(Trivial Name)<br>[CAS Registry Number]<br>InChIKey | $H_s^{cp}$<br>(at $T^\ominus$)<br><br>$\left[\dfrac{\mathrm{mol}}{\mathrm{m}^3\,\mathrm{Pa}}\right]$ | $\dfrac{\mathrm{d}\ln H_s^{cp}}{\mathrm{d}(1/T)}$<br><br>[K] | Reference | Type | Note |
|---|---|---|---|---|---|
| 2,3-dimethylbenzaldehyde<br>$C_9H_{10}O$<br>[5779-93-1]<br>UIFVCPMLQXKEEU-UHFFFAOYSA-N | 1.2<br>2.1<br>$7.6\times10^{-1}$ | | Wang et al. (2017)<br>Wang et al. (2017)<br>Wang et al. (2017) | Q<br>Q<br>Q | 80, 238<br>80, 239<br>80, 240 |
| 3,4-dimethylbenzaldehyde<br>$C_9H_{10}O$<br>[5973-71-7]<br>POQJHLBMLVTHAU-UHFFFAOYSA-N | 1.2<br>1.5<br>3.8 | | Wang et al. (2017)<br>Wang et al. (2017)<br>Wang et al. (2017) | Q<br>Q<br>Q | 80, 238<br>80, 239<br>80, 240 |
| 3,5-dimethylbenzaldehyde<br>$C_9H_{10}O$<br>[5779-95-3]<br>NBEFMISJJNGCIZ-UHFFFAOYSA-N | 1.2<br>$9.6\times10^{-1}$<br>2.0 | | Wang et al. (2017)<br>Wang et al. (2017)<br>Wang et al. (2017) | Q<br>Q<br>Q | 80, 238<br>80, 239<br>80, 240 |
| 2-naphthaldehyde<br>$C_{11}H_8O$<br>[66-99-9]<br>PJKVFARRVXDXAD-UHFFFAOYSA-N | $6.0\times10^1$ | | Abraham et al. (2019) | Q | |
| cyclemax<br>$C_{12}H_{16}O$<br>(4-(1-methylethyl)-<br>benzenepropanal)<br>[7775-00-0]<br>RLEFOSDUWZYGOS-UHFFFAOYSA-N | $7.8\times10^{-1}$ | | Dupeux et al. (2022) | Q | 259 |
| bourgeonal<br>$C_{13}H_{18}O$<br>[18127-01-0]<br>FZJUFJKVIYFBSY-UHFFFAOYSA-N | 1.1 | | Dupeux et al. (2022) | Q | 259 |
| florhydral<br>$C_{13}H_{18}O$<br>[125109-85-5]<br>OHRBQTOZYGEWCJ-UHFFFAOYSA-N | $4.9\times10^{-1}$ | | Dupeux et al. (2022) | Q | 259 |
| cyclamen aldehyde<br>$C_{13}H_{18}O$<br>[103-95-7]<br>ZFNVDHOSLNRHNN-UHFFFAOYSA-N | $2.3\times10^{-1}$ | | Dupeux et al. (2022) | Q | 259 |
| mimosal<br>$C_{13}H_{16}O$<br>(4-methyl-5-(4-methylphenyl)pent-<br>4-enal)<br>[1226911-69-8]<br>LBKHGAIELUNYML-ZRDIBKRKSA-N | 4.9 | | Dupeux et al. (2022) | Q | 259 |



Table A3.5: Aldehydes (RCHO) (...continued)

| Substance Formula (Trivial Name) [CAS Registry Number] InChIKey | $H_s^{cp}$ (at $T^\ominus$) $\left[\dfrac{\mathrm{mol}}{\mathrm{m^3\,Pa}}\right]$ | $\dfrac{\mathrm{d}\ln H_s^{cp}}{\mathrm{d}(1/T)}$ [K] | Reference | Type | Note |
|---|---|---|---|---|---|
| lilial $C_{14}H_{20}O$ [80-54-6] SDQFDHOLCGWZPU-UHFFFAOYSA-N | $5.7\times10^{-1}$ | | Dupeux et al. (2022) | Q | 259 |
| hivernal $C_{14}H_{18}O$ [300371-33-9] GEPCDOWRWODJEY-UHFFFAOYSA-N | 2.2 | | Dupeux et al. (2022) | Q | 259 |
| silvial $C_{14}H_{20}O$ [6658-48-6] YLIXVKUWWOQREC-UHFFFAOYSA-N | $2.9\times10^{-1}$ | | Dupeux et al. (2022) | Q | 259 |
| mefloral $C_{14}H_{20}O$ [62518-65-4] GLZRHVTZLDNUQP-UHFFFAOYSA-N | $3.2\times10^{-1}$ | | Dupeux et al. (2022) | Q | 259 |
| nympheal $C_{14}H_{20}O$ (3-(4-isobutyl-2-methylphenyl)propanal) [1637294-12-2] UKZXPOJABTXLMK-UHFFFAOYSA-N | $8.2\times10^{-1}$ | | Dupeux et al. (2022) | Q | 259 |
| hexyl cinnamic aldehyde $C_{15}H_{20}O$ [101-86-0] GUUHFMWKWLOQMM-UHFFFAOYSA-N | 1.0 | | Dupeux et al. (2022) | Q | 259 |
| $\alpha$-amyl cinnamaldehyde $C_{14}H_{18}O$ [122-40-7] HMKKIXGYKWDQSV-KAMYIIQDSA-N | 1.3 | | HSDB (2015) | Q | 447 |
| MCM:HCOCH2CO3H $C_3H_4O_4$ CFPRXLWZTODTLN-UHFFFAOYSA-N | $8.0\times10^4$ $3.6\times10^3$ $3.2\times10^1$ | | Wang et al. (2017) Wang et al. (2017) Wang et al. (2017) | Q Q Q | 80, 238 80, 239 80, 240 |
| MCM:PROPALOOH $C_3H_6O_3$ UTLVMFCPERHVKU-UHFFFAOYSA-N | $6.9\times10^3$ $8.7\times10^1$ $3.6\times10^1$ | | Wang et al. (2017) Wang et al. (2017) Wang et al. (2017) | Q Q Q | 80, 238 80, 239 80, 240 |
| MCM:BUTALAOOH $C_4H_8O_3$ UBMPIIYSSVMXGT-UHFFFAOYSA-N | $5.5\times10^3$ $4.6\times10^1$ $1.2\times10^1$ | | Wang et al. (2017) Wang et al. (2017) Wang et al. (2017) | Q Q Q | 80, 238 80, 239 80, 240 |
| MCM:BUTALO2H $C_4H_8O_3$ HFNSAVKBHMZYBR-UHFFFAOYSA-N | $5.5\times10^3$ $2.7\times10^3$ $8.7\times10^2$ | | Wang et al. (2017) Wang et al. (2017) Wang et al. (2017) | Q Q Q | 80, 238 80, 239 80, 240 |



Table A3.5: Aldehydes (RCHO) (...continued)

| Substance Formula (Trivial Name) [CAS Registry Number] InChIKey | $H_s^{cp}$ (at $T^\ominus$) $\left[\dfrac{\mathrm{mol}}{\mathrm{m^3\,Pa}}\right]$ | $\dfrac{\mathrm{d}\ln H_s^{cp}}{\mathrm{d}(1/T)}$ [K] | Reference | Type | Note |
|---|---|---|---|---|---|
| MCM:C3MDIALOOH | $3.0\times10^6$ | | Wang et al. (2017) | Q | 80, 238 |
| $C_4H_6O_4$ | $4.7\times10^3$ | | Wang et al. (2017) | Q | 80, 239 |
| WJWKLOFSCXWKQD-UHFFFAOYSA-N | 7.3 | | Wang et al. (2017) | Q | 80, 240 |
| MCM:CHOC2CO3H | $7.1\times10^4$ | | Wang et al. (2017) | Q | 80, 238 |
| $C_4H_6O_4$ | $1.1\times10^4$ | | Wang et al. (2017) | Q | 80, 239 |
| ZCKJJZIUFVHIST-UHFFFAOYSA-N | $1.9\times10^2$ | | Wang et al. (2017) | Q | 80, 240 |
| MCM:IBUTALBO2H | $5.5\times10^3$ | | Wang et al. (2017) | Q | 80, 238 |
| $C_4H_8O_3$ | $3.6\times10^3$ | | Wang et al. (2017) | Q | 80, 239 |
| OSYMKIVGDGNGKG-UHFFFAOYSA-N | $5.4\times10^2$ | | Wang et al. (2017) | Q | 80, 240 |
| MCM:IBUTALO2H | $3.8\times10^3$ | | Wang et al. (2017) | Q | 80, 238 |
| $C_4H_8O_3$ | $1.8\times10^1$ | | Wang et al. (2017) | Q | 80, 239 |
| UINKREZTGVLFAU-UHFFFAOYSA-N | $2.6\times10^1$ | | Wang et al. (2017) | Q | 80, 240 |
| MCM:MALDALCO3H | $2.9\times10^5$ | | Wang et al. (2017) | Q | 80, 238 |
| $C_4H_4O_4$ | $3.5\times10^4$ | | Wang et al. (2017) | Q | 80, 239 |
| QFBRACYMKXGFRM-UHFFFAOYSA-N | $3.7\times10^1$ | | Wang et al. (2017) | Q | 80, 240 |
| MCM:PRPAL2CO3H | $7.1\times10^4$ | | Wang et al. (2017) | Q | 80, 238 |
| $C_4H_6O_4$ | $2.3\times10^3$ | | Wang et al. (2017) | Q | 80, 239 |
| IGHWWGZEWMXWCB-UHFFFAOYSA-N | 8.0 | | Wang et al. (2017) | Q | 80, 240 |
| MCM:C3EDIALOOH | $2.7\times10^6$ | | Wang et al. (2017) | Q | 80, 238 |
| $C_5H_8O_4$ | $2.1\times10^3$ | | Wang et al. (2017) | Q | 80, 239 |
| IOKXWRCVGKGMQG-UHFFFAOYSA-N | 4.9 | | Wang et al. (2017) | Q | 80, 240 |
| MCM:C3M3CHOOOH | $3.1\times10^3$ | | Wang et al. (2017) | Q | 80, 238 |
| $C_5H_{10}O_3$ | $8.7\times10^2$ | | Wang et al. (2017) | Q | 80, 239 |
| FJYKIGXUSFRBER-UHFFFAOYSA-N | $3.1\times10^2$ | | Wang et al. (2017) | Q | 80, 240 |
| MCM:C4CHOAOOH | $4.5\times10^3$ | | Wang et al. (2017) | Q | 80, 238 |
| $C_5H_{10}O_3$ | $3.0\times10^1$ | | Wang et al. (2017) | Q | 80, 239 |
| ZCMMOPXACREGKB-UHFFFAOYSA-N | 6.3 | | Wang et al. (2017) | Q | 80, 240 |
| MCM:C4CHOBOOH | $4.5\times10^3$ | | Wang et al. (2017) | Q | 80, 238 |
| $C_5H_{10}O_3$ | $1.5\times10^3$ | | Wang et al. (2017) | Q | 80, 239 |
| JELNOURJPXUYRE-UHFFFAOYSA-N | $3.0\times10^2$ | | Wang et al. (2017) | Q | 80, 240 |
| MCM:C4CODBCO3H | $2.0\times10^5$ | | Wang et al. (2017) | Q | 80, 238 |
| $C_5H_6O_4$ | $4.2\times10^4$ | | Wang et al. (2017) | Q | 80, 239 |
| JLZDPUBRQDSTHL-UHFFFAOYSA-N | $6.9\times10^{-1}$ | | Wang et al. (2017) | Q | 80, 240 |
| MCM:C514OOH | $4.1\times10^6$ | | Wang et al. (2017) | Q | 80, 238 |
| $C_5H_8O_4$ | $2.0\times10^6$ | | Wang et al. (2017) | Q | 80, 239 |
| PUFIIBDSTKVRQR-UHFFFAOYSA-N | $6.6\times10^4$ | | Wang et al. (2017) | Q | 80, 240 |
| MCM:C5DIALOOH | $1.6\times10^7$ | | Wang et al. (2017) | Q | 80, 238 |
| $C_5H_6O_4$ | $7.3\times10^4$ | | Wang et al. (2017) | Q | 80, 239 |
| MJFLAVCETRHPDB-UHFFFAOYSA-N | $3.0\times10^4$ | | Wang et al. (2017) | Q | 80, 240 |





Table A3.5: Aldehydes (RCHO) (... continued)

| Substance Formula (Trivial Name) [CAS Registry Number] InChIKey | $H_s^{cp}$ (at $T^{\ominus}$) $\left[\dfrac{\mathrm{mol}}{\mathrm{m^3\,Pa}}\right]$ | $\dfrac{\mathrm{d}\ln H_s^{cp}}{\mathrm{d}(1/T)}$ [K] | Reference | Type | Note |
|---|---|---|---|---|---|
| MCM:CHOC4OOH | $4.3\times10^3$ | | Wang et al. (2017) | Q | 80, 238 |
| $C_5H_{10}O_3$ | $5.3\times10^3$ | | Wang et al. (2017) | Q | 80, 239 |
| CTZAXNMAMSERLK-UHFFFAOYSA-N | $2.0\times10^2$ | | Wang et al. (2017) | Q | 80, 240 |
| MCM:CO1M22CO3H | $4.1\times10^4$ | | Wang et al. (2017) | Q | 80, 238 |
| $C_5H_8O_4$ | $8.0\times10^2$ | | Wang et al. (2017) | Q | 80, 239 |
| HHOOHNHNSGMLRV-UHFFFAOYSA-N | 8.9 | | Wang et al. (2017) | Q | 80, 240 |
| MCM:IC4CHOAOOH | $5.1\times10^3$ | | Wang et al. (2017) | Q | 80, 238 |
| $C_5H_{10}O_3$ | $3.8\times10^1$ | | Wang et al. (2017) | Q | 80, 239 |
| MEJCTHPUJKQILI-UHFFFAOYSA-N | 8.3 | | Wang et al. (2017) | Q | 80, 240 |
| MCM:MC3ODBCO3H | $2.0\times10^5$ | | Wang et al. (2017) | Q | 80, 238 |
| $C_5H_6O_4$ | $3.9\times10^4$ | | Wang et al. (2017) | Q | 80, 239 |
| UTZOYJZISPOIDM-UHFFFAOYSA-N | $3.9\times10^1$ | | Wang et al. (2017) | Q | 80, 240 |
| MCM:C522CO3H | $1.1\times10^5$ | | Wang et al. (2017) | Q | 80, 238 |
| $C_6H_8O_4$ | $8.0\times10^3$ | | Wang et al. (2017) | Q | 80, 239 |
| NCACXMVSAKPDND-UHFFFAOYSA-N | $1.2\times10^1$ | | Wang et al. (2017) | Q | 80, 240 |
| MCM:C615OOH | $2.5\times10^6$ | | Wang et al. (2017) | Q | 80, 238 |
| $C_6H_{10}O_4$ | $2.2\times10^4$ | | Wang et al. (2017) | Q | 80, 239 |
| XHCYBISQWAKVQU-UHFFFAOYSA-N | $3.2\times10^2$ | | Wang et al. (2017) | Q | 80, 240 |
| MCM:C6DIALOOH | $3.2\times10^6$ | | Wang et al. (2017) | Q | 80, 238 |
| $C_6H_{10}O_4$ | $1.5\times10^6$ | | Wang et al. (2017) | Q | 80, 239 |
| UGGYJIUZOSPDSO-UHFFFAOYSA-N | $5.4\times10^4$ | | Wang et al. (2017) | Q | 80, 240 |
| MCM:CHOC4CO3H | $4.6\times10^4$ | | Wang et al. (2017) | Q | 80, 238 |
| $C_6H_{10}O_4$ | $3.7\times10^3$ | | Wang et al. (2017) | Q | 80, 239 |
| RHHSYYPMFOTTRH-UHFFFAOYSA-N | $6.8\times10^1$ | | Wang et al. (2017) | Q | 80, 240 |
| MCM:CO1C6OOH | $3.5\times10^3$ | | Wang et al. (2017) | Q | 80, 238 |
| $C_6H_{12}O_3$ | $3.4\times10^3$ | | Wang et al. (2017) | Q | 80, 239 |
| HEPNQRDDKATRCQ-UHFFFAOYSA-N | $8.0\times10^2$ | | Wang et al. (2017) | Q | 80, 240 |
| MCM:C615CO3H | $3.0\times10^7$ | | Wang et al. (2017) | Q | 80, 238 |
| $C_7H_{10}O_5$ | $3.4\times10^5$ | | Wang et al. (2017) | Q | 80, 239 |
| DXQJLRIPGQOCBO-UHFFFAOYSA-N | $1.2\times10^2$ | | Wang et al. (2017) | Q | 80, 240 |
| MCM:C729OOH | $7.6\times10^3$ | | Wang et al. (2017) | Q | 80, 238 |
| $C_7H_{12}O_3$ | $2.6\times10^3$ | | Wang et al. (2017) | Q | 80, 239 |
| PQESPNYRBXTFKD-UHFFFAOYSA-N | $4.6\times10^1$ | | Wang et al. (2017) | Q | 80, 240 |
| MCM:C729CO3H | $7.8\times10^4$ | | Wang et al. (2017) | Q | 80, 238 |
| $C_8H_{12}O_4$ | $1.6\times10^3$ | | Wang et al. (2017) | Q | 80, 239 |
| MJXLYBJUYDBXAC-UHFFFAOYSA-N | $2.5\times10^1$ | | Wang et al. (2017) | Q | 80, 240 |
| MCM:C810OOH | $1.6\times10^6$ | | Wang et al. (2017) | Q | 80, 238 |
| $C_8H_{14}O_4$ | $5.5\times10^5$ | | Wang et al. (2017) | Q | 80, 239 |
| YJCLQCSGAKCWEJ-UHFFFAOYSA-N | $1.6\times10^4$ | | Wang et al. (2017) | Q | 80, 240 |





Table A3.5: Aldehydes (RCHO) (. . . continued)

| Substance Formula (Trivial Name) [CAS Registry Number] InChIKey | $H_s^{cp}$ (at $T^{\ominus}$) $\left[\dfrac{\mathrm{mol}}{\mathrm{m}^3\,\mathrm{Pa}}\right]$ | $\dfrac{\mathrm{d}\ln H_s^{cp}}{\mathrm{d}(1/T)}$ [K] | Reference | Type | Note |
|---|---|---|---|---|---|
| MCM:C822OOH $C_8H_{14}O_3$ XWYULULFGKWPMD-UHFFFAOYSA-N | $8.8\times10^3$ $6.0\times10^3$ $2.8\times10^3$ $1.1\times10^2$ | 11000 | Wieser et al. (2023) Wang et al. (2017) Wang et al. (2017) Wang et al. (2017) | Q Q Q Q | 437 80, 238 80, 239 80, 240 |
| MCM:C830OOH $C_8H_{14}O_3$ QPQAMUJVCCDVQB-UHFFFAOYSA-N | $6.5\times10^3$ $4.4\times10^3$ $1.4\times10^2$ | | Wang et al. (2017) Wang et al. (2017) Wang et al. (2017) | Q Q Q | 80, 238 80, 239 80, 240 |
| MCM:C831OOH $C_8H_{14}O_4$ IGXVNESWPQHYEP-UHFFFAOYSA-N | $1.6\times10^6$ $3.0\times10^5$ $3.0\times10^4$ | | Wang et al. (2017) Wang et al. (2017) Wang et al. (2017) | Q Q Q | 80, 238 80, 239 80, 240 |
| MCM:C89OOH $C_8H_{14}O_3$ TYPKFPBYWMWINI-UHFFFAOYSA-N | $6.5\times10^3$ $4.4\times10^3$ $8.0\times10^3$ | | Wang et al. (2017) Wang et al. (2017) Wang et al. (2017) | Q Q Q | 80, 238 80, 239 80, 240 |
| MCM:C822CO3H $C_9H_{14}O_4$ DRDXOQWGQPPQFG-UHFFFAOYSA-N | $1.7\times10^4$ $7.3\times10^4$ $1.1\times10^3$ $3.8\times10^1$ | 14000 | Wieser et al. (2023) Wang et al. (2017) Wang et al. (2017) Wang et al. (2017) | Q Q Q Q | 437 80, 238 80, 239 80, 240 |
| MCM:C830CO3H $C_9H_{14}O_4$ YIGLTTCQERBPNX-UHFFFAOYSA-N | $7.8\times10^4$ $2.8\times10^3$ $3.6\times10^1$ | | Wang et al. (2017) Wang et al. (2017) Wang et al. (2017) | Q Q Q | 80, 238 80, 239 80, 240 |
| MCM:C89CO3H $C_9H_{14}O_4$ QJRQWHYQDGSSTJ-UHFFFAOYSA-N | $7.8\times10^4$ $2.9\times10^3$ $1.0\times10^2$ | | Wang et al. (2017) Wang et al. (2017) Wang et al. (2017) | Q Q Q | 80, 238 80, 239 80, 240 |
| MCM:C126OOH $C_{12}H_{20}O_3$ KEVCPNVRIXMZTI-UHFFFAOYSA-N | $7.1\times10^3$ $4.9\times10^3$ $1.8\times10^2$ | | Wang et al. (2017) Wang et al. (2017) Wang et al. (2017) | Q Q Q | 80, 238 80, 239 80, 240 |
| MCM:C126CO3H $C_{13}H_{20}O_4$ OZAZBNTYEOZXRC-UHFFFAOYSA-N | $8.5\times10^4$ $1.9\times10^3$ $2.2\times10^2$ | | Wang et al. (2017) Wang et al. (2017) Wang et al. (2017) | Q Q Q | 80, 238 80, 239 80, 240 |
| MCM:C136OOH $C_{13}H_{22}O_3$ MAMVWZJYPZZMSJ-UHFFFAOYSA-N | $5.8\times10^3$ $4.3\times10^3$ $4.1\times10^2$ | | Wang et al. (2017) Wang et al. (2017) Wang et al. (2017) | Q Q Q | 80, 238 80, 239 80, 240 |
| MCM:C136CO3H $C_{14}H_{22}O_4$ JKRMUSIUVDAEPW-UHFFFAOYSA-N | $6.8\times10^4$ $1.4\times10^3$ $7.8\times10^2$ | | Wang et al. (2017) Wang et al. (2017) Wang et al. (2017) | Q Q Q | 80, 238 80, 239 80, 240 |
| MCM:HCOCOHCO3H $C_3H_4O_5$ DIZMXPJJIVKAJT-UHFFFAOYSA-N | $2.9\times10^6$ $1.3\times10^4$ $6.5\times10^2$ | | Wang et al. (2017) Wang et al. (2017) Wang et al. (2017) | Q Q Q | 80, 238 80, 239 80, 240 |
| MCM:HOCHOCOOH $C_3H_6O_4$ GWWSSJVARZQKFN-UHFFFAOYSA-N | $2.4\times10^7$ $1.3\times10^5$ $2.0\times10^3$ | | Wang et al. (2017) Wang et al. (2017) Wang et al. (2017) | Q Q Q | 80, 238 80, 239 80, 240 |





Table A3.5: Aldehydes (RCHO) (. . . continued)

| Substance Formula (Trivial Name) [CAS Registry Number] InChIKey | $H_s^{cp}$ (at $T^{\ominus}$) $\left[\dfrac{\text{mol}}{\text{m}^3\,\text{Pa}}\right]$ | $\dfrac{\mathrm{d}\ln H_s^{cp}}{\mathrm{d}(1/T)}$ [K] | Reference | Type | Note |
|---|---|---|---|---|---|
| MCM:OCCOHCOOH | $1.2\times10^6$ | | Wang et al. (2017) | Q | 80, 238 |
| $C_3H_6O_4$ | $5.9\times10^4$ | | Wang et al. (2017) | Q | 80, 239 |
| UFAXCNDNIIDPQE-UHFFFAOYSA-N | $2.6\times10^4$ | | Wang et al. (2017) | Q | 80, 240 |
| MCM:C4OCCOHOOH | $1.1\times10^6$ | | Wang et al. (2017) | Q | 80, 238 |
| $C_4H_8O_4$ | $1.2\times10^5$ | | Wang et al. (2017) | Q | 80, 239 |
| JJIHBDXOHFOUDV-UHFFFAOYSA-N | $7.8\times10^3$ | | Wang et al. (2017) | Q | 80, 240 |
| MCM:COCCOHCOOH | $2.2\times10^7$ | | Wang et al. (2017) | Q | 80, 238 |
| $C_4H_8O_4$ | $2.0\times10^5$ | | Wang et al. (2017) | Q | 80, 239 |
| XTEYRSCVGRLIOG-UHFFFAOYSA-N | $8.1\times10^2$ | | Wang et al. (2017) | Q | 80, 240 |
| MCM:COHM2CO3H | $1.7\times10^6$ | | Wang et al. (2017) | Q | 80, 238 |
| $C_4H_6O_5$ | $4.6\times10^3$ | | Wang et al. (2017) | Q | 80, 239 |
| AEAZUHMXWJBPCW-UHFFFAOYSA-N | $2.8\times10^1$ | | Wang et al. (2017) | Q | 80, 240 |
| MCM:HMACROOH | $3.9\times10^{10}$ | | Wang et al. (2017) | Q | 80, 238 |
| $C_4H_8O_5$ | $2.4\times10^8$ | | Wang et al. (2017) | Q | 80, 239 |
| AWITZWJPGUOCMB-UHFFFAOYSA-N | $4.6\times10^4$ | | Wang et al. (2017) | Q | 80, 240 |
| MCM:MACROHOOH | $6.6\times10^5$ | | Wang et al. (2017) | Q | 80, 238 |
| $C_4H_8O_4$ | $7.1\times10^4$ | | Wang et al. (2017) | Q | 80, 239 |
| HWQXXVVDRNBYQB-UHFFFAOYSA-N | $6.2\times10^3$ | | Wang et al. (2017) | Q | 80, 240 |
| MCM:MACROOH | $1.3\times10^7$ | | Wang et al. (2017) | Q | 80, 238 |
| $C_4H_8O_4$ | $7.6\times10^4$ | | Wang et al. (2017) | Q | 80, 239 |
| MOSGWPXPTZVSII-UHFFFAOYSA-N | $5.8\times10^2$ | | Wang et al. (2017) | Q | 80, 240 |
| MCM:MALDIALOOH | $1.0\times10^9$ | | Wang et al. (2017) | Q | 80, 238 |
| $C_4H_6O_5$ | $6.2\times10^6$ | | Wang et al. (2017) | Q | 80, 239 |
| MNYHDFHIZDEDLP-UHFFFAOYSA-N | $3.6\times10^3$ | | Wang et al. (2017) | Q | 80, 240 |
| MCM:ALC4DOLOOH | $1.6\times10^{10}$ | | Wang et al. (2017) | Q | 80, 238 |
| $C_5H_{10}O_5$ | $3.8\times10^8$ | | Wang et al. (2017) | Q | 80, 239 |
| BTJWWKUWEVGVGA-UHFFFAOYSA-N | $5.0\times10^6$ | | Wang et al. (2017) | Q | 80, 240 |
| MCM:C4MALOHOOH | $5.6\times10^8$ | | Wang et al. (2017) | Q | 80, 238 |
| $C_5H_8O_5$ | $6.6\times10^6$ | | Wang et al. (2017) | Q | 80, 239 |
| NJMKFLHNNBFYIH-UHFFFAOYSA-N | $3.6\times10^2$ | | Wang et al. (2017) | Q | 80, 240 |
| MCM:C57OOH | $1.3\times10^{10}$ | | Wang et al. (2017) | Q | 80, 238 |
| $C_5H_{10}O_5$ | $8.5\times10^7$ | | Wang et al. (2017) | Q | 80, 239 |
| VJGRRPOTERGJKF-UHFFFAOYSA-N | $1.2\times10^4$ | | Wang et al. (2017) | Q | 80, 240 |
| MCM:C58OOH | $1.9\times10^9$ | | Wang et al. (2017) | Q | 80, 238 |
| $C_5H_{10}O_5$ | $3.7\times10^8$ | | Wang et al. (2017) | Q | 80, 239 |
| MTCQYLXAFMKVMT-UHFFFAOYSA-N | $3.1\times10^4$ | | Wang et al. (2017) | Q | 80, 240 |
| MCM:CHOC4OHOOH | $1.5\times10^7$ | | Wang et al. (2017) | Q | 80, 238 |
| $C_5H_{10}O_4$ | $1.4\times10^6$ | | Wang et al. (2017) | Q | 80, 239 |
| LTZYUHWRVSINRM-UHFFFAOYSA-N | $1.6\times10^5$ | | Wang et al. (2017) | Q | 80, 240 |





Table A3.5: Aldehydes (RCHO) (...continued)

| Substance Formula (Trivial Name) [CAS Registry Number] InChIKey | $H_s^{cp}$ (at $T^\ominus$) $\left[\dfrac{\text{mol}}{\text{m}^3\,\text{Pa}}\right]$ | $\dfrac{\mathrm{d}\ln H_s^{cp}}{\mathrm{d}(1/T)}$ [K] | Reference | Type | Note |
|---|---|---|---|---|---|
| MCM:HIEB1OOH $C_5H_{10}O_6$ COFCBVJXHDFSHS-UHFFFAOYSA-N | $3.7\times10^{13}$ $1.6\times10^{11}$ $6.5\times10^{6}$ | | Wang et al. (2017) Wang et al. (2017) Wang et al. (2017) | Q Q Q | 80, 238 80, 239 80, 240 |
| MCM:HIEB2OOH $C_5H_{10}O_6$ PKIUCWJPHHVKPZ-UHFFFAOYSA-N | $2.4\times10^{13}$ $1.0\times10^{11}$ $1.2\times10^{7}$ | | Wang et al. (2017) Wang et al. (2017) Wang et al. (2017) | Q Q Q | 80, 238 80, 239 80, 240 |
| MCM:HOC4CHOOOH $C_5H_{10}O_4$ LRYOXTYJCQGVIC-UHFFFAOYSA-N | $1.4\times10^{7}$ $6.9\times10^{5}$ $1.9\times10^{4}$ | | Wang et al. (2017) Wang et al. (2017) Wang et al. (2017) | Q Q Q | 80, 238 80, 239 80, 240 |
| MCM:IEB2OOH $C_5H_{10}O_5$ MCOKLECUTPXNOW-UHFFFAOYSA-N | $1.3\times10^{10}$ $3.7\times10^{8}$ $2.7\times10^{4}$ | | Wang et al. (2017) Wang et al. (2017) Wang et al. (2017) | Q Q Q | 80, 238 80, 239 80, 240 |
| MCM:C1H4C5CO3H $C_6H_{10}O_5$ DOEHHISPDHWTCR-UHFFFAOYSA-N | $1.6\times10^{8}$ $5.9\times10^{7}$ $2.4\times10^{3}$ | | Wang et al. (2017) Wang et al. (2017) Wang et al. (2017) | Q Q Q | 80, 238 80, 239 80, 240 |
| MCM:C623OOH $C_6H_{12}O_5$ OGBKQGQRAMSXTF-UHFFFAOYSA-N | $2.8\times10^{10}$ $7.4\times10^{8}$ $1.0\times10^{6}$ | | Wang et al. (2017) Wang et al. (2017) Wang et al. (2017) | Q Q Q | 80, 238 80, 239 80, 240 |
| MCM:C67OOH $C_6H_{12}O_4$ ONTGLCVTLGTTQS-UHFFFAOYSA-N | $1.0\times10^{7}$ $4.2\times10^{5}$ $9.8\times10^{2}$ | | Wang et al. (2017) Wang et al. (2017) Wang et al. (2017) | Q Q Q | 80, 238 80, 239 80, 240 |
| MCM:C68OOH $C_6H_{12}O_4$ ITDKDNQPTMKBIM-UHFFFAOYSA-N | $1.0\times10^{7}$ $4.6\times10^{5}$ $3.4\times10^{2}$ | | Wang et al. (2017) Wang et al. (2017) Wang et al. (2017) | Q Q Q | 80, 238 80, 239 80, 240 |
| MCM:CO1H63OOH $C_6H_{12}O_4$ KJWILEAANBSBHN-UHFFFAOYSA-N | $1.1\times10^{7}$ $1.8\times10^{7}$ $3.8\times10^{5}$ | | Wang et al. (2017) Wang et al. (2017) Wang et al. (2017) | Q Q Q | 80, 238 80, 239 80, 240 |
| MCM:C728OOH $C_7H_{14}O_5$ WJYXJQPIZCIBJO-UHFFFAOYSA-N | $5.3\times10^{10}$ $2.1\times10^{10}$ $1.7\times10^{9}$ $1.4\times10^{7}$ | 18000 | Wieser et al. (2023) Wang et al. (2017) Wang et al. (2017) Wang et al. (2017) | Q Q Q Q | 437 80, 238 80, 239 80, 240 |
| MCM:C730OOH $C_7H_{14}O_5$ YXKHJQXHKYJCPI-UHFFFAOYSA-N | $6.1\times10^{10}$ $2.1\times10^{10}$ $1.1\times10^{10}$ $2.6\times10^{6}$ | 18000 | Wieser et al. (2023) Wang et al. (2017) Wang et al. (2017) Wang et al. (2017) | Q Q Q Q | 437 80, 238 80, 239 80, 240 |
| MCM:C824OOH $C_8H_{14}O_4$ FGEKWAMEWUMQIL-UHFFFAOYSA-N | $8.4\times10^{6}$ $1.9\times10^{7}$ $6.3\times10^{5}$ $6.3\times10^{3}$ | 15000 | Wieser et al. (2023) Wang et al. (2017) Wang et al. (2017) Wang et al. (2017) | Q Q Q Q | 437 80, 238 80, 239 80, 240 |





Table A3.5: Aldehydes (RCHO) (... continued)

| Substance Formula (Trivial Name) [CAS Registry Number] InChIKey | $H_s^{cp}$ (at $T^{\ominus}$) $\left[\dfrac{\text{mol}}{\text{m}^3\,\text{Pa}}\right]$ | $\dfrac{\text{d}\ln H_s^{cp}}{\text{d}(1/T)}$ [K] | Reference | Type | Note |
|---|---|---|---|---|---|
| MCM:C826OOH | $6.3 \times 10^9$ | | Wang et al. (2017) | Q | 80, 238 |
| $C_8H_{14}O_5$ | $5.1 \times 10^8$ | | Wang et al. (2017) | Q | 80, 239 |
| MATCTRFTDTXPPO-UHFFFAOYSA-N | $1.7 \times 10^5$ | | Wang et al. (2017) | Q | 80, 240 |
| MCM:C127OOH | $1.3 \times 10^7$ | | Wang et al. (2017) | Q | 80, 238 |
| $C_{12}H_{20}O_4$ | $1.9 \times 10^7$ | | Wang et al. (2017) | Q | 80, 239 |
| CZSOPCQUOGQYQQ-UHFFFAOYSA-N | $2.3 \times 10^5$ | | Wang et al. (2017) | Q | 80, 240 |
| MCM:C1311OOH | $1.2 \times 10^7$ | | Wang et al. (2017) | Q | 80, 238 |
| $C_{13}H_{22}O_4$ | $1.6 \times 10^7$ | | Wang et al. (2017) | Q | 80, 239 |
| GNGYVQPBVCXXTM-UHFFFAOYSA-N | $1.2 \times 10^6$ | | Wang et al. (2017) | Q | 80, 240 |
| MCM:C1313OOH | $6.0 \times 10^9$ | | Wang et al. (2017) | Q | 80, 238 |
| $C_{13}H_{22}O_5$ | $9.1 \times 10^8$ | | Wang et al. (2017) | Q | 80, 239 |
| ALNYAIDMJTUDIF-UHFFFAOYSA-N | $1.9 \times 10^6$ | | Wang et al. (2017) | Q | 80, 240 |





### A3.6 Ketones (RCOR)

Table A3.6: Ketones (RCOR)

| Substance Formula (Trivial Name) [CAS Registry Number] InChIKey | $H_s^{cp}$ (at $T^{\ominus}$) $\left[\dfrac{\text{mol}}{\text{m}^3\,\text{Pa}}\right]$ | $\dfrac{\text{d}\ln H_s^{cp}}{\text{d}(1/T)}$ [K] | Reference | Type | Note |
|---|---|---|---|---|---|
| propanone | $2.7\times10^{-1}$ | 5500 | Burkholder et al. (2019) | L | |
| $CH_3COCH_3$ | $2.7\times10^{-1}$ | 5500 | Burkholder et al. (2015) | L | |
| (acetone) | $2.9\times10^{-1}$ | 5300 | Brockbank (2013) | L | 1, 476 |
| [67-64-1] | $2.7\times10^{-1}$ | 5500 | Sander et al. (2011) | L | |
| CSCPPACGZOOCGX-UHFFFAOYSA-N | $3.3\times10^{-1}$ | 5300 | Poulain et al. (2010) | L | |
| | $2.8\times10^{-1}$ | 5100 | Sander et al. (2006) | L | |
| | $2.6\times10^{-1}$ | 5700 | Fogg and Sangster (2003) | L | |
| | $2.8\times10^{-1}$ | 4800 | Staudinger and Roberts (2001) | L | |
| | $2.7\times10^{-1}$ | 5000 | Plyasunov and Shock (2001) | L | |
| | $3.0\times10^{-1}$ | 4600 | Staudinger and Roberts (1996) | L | |
| | $2.9\times10^{-1}$ | 5100 | Poulain et al. (2010) | M | |
| | $3.3\times10^{-1}$ | 4500 | O'Farrell and Waghorne (2010) | M | |
| | $2.6\times10^{-1}$ | 5400 | Ji and Evans (2007) | M | |
| | $2.4\times10^{-1}$ | 4200 | Falabella et al. (2006) | M | 11, 338 |
| | $2.6\times10^{-1}$ | 6400 | Strekowski and George (2005) | M | |
| | $2.4\times10^{-1}$ | | Straver and de Loos (2005) | M | |
| | $2.4\times10^{-1}$ | 4300 | Chai et al. (2005) | M | 11 |
| | $2.7\times10^{-1}$ | | Nozière and Riemer (2003) | M | 79 |
| | $1.0\times10^{-1}$ | | Ayuttaya et al. (2001) | M | 340 |
| | $9.4\times10^{-4}$ | | Ayuttaya et al. (2001) | M | 341 |
| | $5.3\times10^{-1}$ | | Ayuttaya et al. (2001) | M | 342 |
| | $9.6\times10^{-2}$ | | Welke et al. (1998) | M | |
| | $2.7\times10^{-1}$ | 5300 | Benkelberg et al. (1995) | M | |
| | $2.7\times10^{-1}$ | | Hoff et al. (1993) | M | |
| | $1.7\times10^{-1}$ | | Yu (1992) | M | 12 |
| | $3.2\times10^{-1}$ | 5800 | Betterton (1991) | M | |
| | $3.5\times10^{-1}$ | 3800 | Zhou and Mopper (1990) | M | 456 |
| | $1.2\times10^{-1}$ | | Guitart et al. (1989) | M | 14 |
| | $1.4\times10^{-1}$ | | Hellmann (1987) | M | 87 |
| | $2.5\times10^{-1}$ | 4800 | Snider and Dawson (1985) | M | |
| | $3.2\times10^{-1}$ | 5400 | Schoene and Steinhanses (1985) | M | |
| | $1.9\times10^{-1}$ | | Richon et al. (1985) | M | 38 |
| | $2.6\times10^{-1}$ | 5100 | Lichtenbelt and Schram (1985) | M | 477 |
| | $2.0\times10^{-1}$ | 7800 | Ioffe et al. (1984) | M | |
| | $1.5\times10^{-1}$ | | Sato and Nakajima (1979a) | M | 14 |
| | $2.5\times10^{-1}$ | | Vitenberg et al. (1975) | M | |
| | $2.5\times10^{-1}$ | | Vitenberg et al. (1974) | M | |
| | $3.2\times10^{-1}$ | | Vitenberg et al. (1974) | M | |
| | $2.5\times10^{-1}$ | | Buttery et al. (1969) | M | |
| | $3.1\times10^{-1}$ | | Nelson and Hoff (1968) | M | 297 |
| | $2.8\times10^{-1}$ | | Burnett (1963) | M | |
| | $1.8\times10^{-2}$ | | Abraham and Acree (2007) | V | |
| | $2.6\times10^{-1}$ | | Hwang et al. (1992) | V | |
| | $3.1\times10^{-2}$ | 3100 | Djerki and Laub (1988) | V | |
| | $2.4\times10^{-1}$ | | Rathbun and Tai (1982) | V | |





Table A3.6: Ketones (RCOR) (...continued)

| Substance Formula (Trivial Name) [CAS Registry Number] InChIKey | $H_s^{cp}$ (at $T^{\ominus}$) $\left[\dfrac{\mathrm{mol}}{\mathrm{m^3\,Pa}}\right]$ | $\dfrac{\mathrm{d}\ln H_s^{cp}}{\mathrm{d}(1/T)}$ [K] | Reference | Type | Note |
|---|---|---|---|---|---|
| | $3.1\times10^{-2}$ | | Hine and Weimar (1965) | R | |
| | $3.0\times10^{-1}$ | | Butler and Ramchandani (1935) | R | |
| | $2.5\times10^{-1}$ | 4900 | Bagno et al. (1991) | T | 473 |
| | $2.1\times10^{-1}$ | | Yaws (2003) | X | 258 |
| | $2.2\times10^{-1}$ | 5000 | Schaffer and Daubert (1969) | X | 298 |
| | $3.0\times10^{-2}$ | 3300 | Janini and Quaddora (1986) | X | 298 |
| | $3.0\times10^{-1}$ | | Gaffney and Senum (1984) | X | 389 |
| | $2.7\times10^{-1}$ | | Cabani et al. (1981) | C | |
| | $6.1\times10^{-1}$ | | Dupeux et al. (2022) | Q | 259 |
| | $2.6\times10^{-1}$ | | Hayer et al. (2022) | Q | 20 |
| | $1.2\times10^{-1}$ | | Keshavarz et al. (2022) | Q | |
| | $3.2\times10^{-2}$ | | Duchowicz et al. (2020) | Q | |
| | $6.6\times10^{-2}$ | | Wang et al. (2017) | Q | 80, 238 |
| | $4.3\times10^{-1}$ | | Wang et al. (2017) | Q | 80, 239 |
| | $7.1\times10^{-1}$ | | Wang et al. (2017) | Q | 80, 240 |
| | $2.5\times10^{-1}$ | | Li et al. (2014) | Q | 241 |
| | $2.5\times10^{-1}$ | | Raventos-Duran et al. (2010) | Q | 271, 243 |
| | $2.0\times10^{-1}$ | | Raventos-Duran et al. (2010) | Q | 244 |
| | $2.0\times10^{-1}$ | | Raventos-Duran et al. (2010) | Q | 245 |
| | $1.4\times10^{-1}$ | | Hilal et al. (2008) | Q | |
| | $4.0\times10^{-1}$ | | Modarresi et al. (2007) | Q | 67 |
| | | 5500 | Kühne et al. (2005) | Q | |
| | $2.5\times10^{-1}$ | | Yaffe et al. (2003) | Q | 248, 249 |
| | $2.5\times10^{-1}$ | | English and Carroll (2001) | Q | 230, 231 |
| | $2.4\times10^{-2}$ | | Katritzky et al. (1998) | Q | |
| | $2.1\times10^{-1}$ | | Nirmalakhandan et al. (1997) | Q | |
| | $1.9\times10^{-1}$ | | Suzuki et al. (1992) | Q | 232 |
| | $2.5\times10^{-1}$ | | Taft et al. (1985) | Q | |
| | $2.8\times10^{-1}$ | | Duchowicz et al. (2020) | ? | 185, 21 |
| | $2.5\times10^{-1}$ | | Mackay et al. (2006c) | ? | 21 |
| | | 5100 | Kühne et al. (2005) | ? | |
| | $1.5\times10^{-1}$ | | Yaws (1999) | ? | 21 |
| | $1.8\times10^{-1}$ | | Yaws et al. (1998) | ? | |
| | $1.6\times10^{-1}$ | | Abraham and Weathersby (1994) | ? | 21 |
| | $2.3\times10^{-1}$ | | Yaws and Yang (1992) | ? | 21 |
| | $2.5\times10^{-1}$ | | Abraham et al. (1990) | ? | |
| propanone-2-13C CH$_3$COCH$_3$ (acetone-2-13C) [3881-06-9] CSCPPACGZOOCGX-LBPDFUHNSA-N | $3.1\times10^{-1}$ | 5300 | Hiatt (2013) | M | |
| 1-hydroxypropanone CH$_3$COCH$_2$OH (hydroxyacetone) [116-09-6] XLSMFKSTNGKWQX-UHFFFAOYSA-N | $7.7\times10^{1}$ | | Lee and Zhou (1993) | C | 87 |
| | 8.3 | | Wang et al. (2017) | Q | 80, 238 |
| | $1.4\times10^{2}$ | | Wang et al. (2017) | Q | 80, 239 |
| | $1.9\times10^{1}$ | | Wang et al. (2017) | Q | 80, 240 |





Table A3.6: Ketones (RCOR) (... continued)

| Substance<br>Formula<br>(Trivial Name)<br>[CAS Registry Number]<br>InChIKey | $H_s^{cp}$<br>(at $T^{\ominus}$)<br>$\left[\dfrac{\text{mol}}{\text{m}^3\,\text{Pa}}\right]$ | $\dfrac{\text{d}\ln H_s^{cp}}{\text{d}(1/T)}$<br><br>[K] | Reference | Type | Note |
|---|---|---|---|---|---|
| butanone | $1.8\times10^{-1}$ | 5700 | Burkholder et al. (2019) | L | |
| $C_2H_5COCH_3$ | $1.8\times10^{-1}$ | 5700 | Burkholder et al. (2015) | L | |
| (methyl ethyl ketone; MEK) | $1.8\times10^{-1}$ | 5600 | Brockbank (2013) | L | 1 |
| [78-93-3] | $1.8\times10^{-1}$ | 5700 | Sander et al. (2011) | L | |
| ZWEHNKRNPOVVGH-UHFFFAOYSA-N | $1.8\times10^{-1}$ | 5700 | Sander et al. (2006) | L | |
| | $1.9\times10^{-1}$ | 4600 | Fogg and Sangster (2003) | L | |
| | $1.8\times10^{-1}$ | 5400 | Staudinger and Roberts (2001) | L | |
| | $1.8\times10^{-1}$ | 5500 | Plyasunov and Shock (2001) | L | |
| | $2.0\times10^{-1}$ | 5000 | Staudinger and Roberts (1996) | L | |
| | $1.0\times10^{-1}$ | | Kim and Kim (2014) | M | |
| | $9.5\times10^{-2}$ | | Helburn et al. (2008) | M | |
| | $2.1\times10^{-1}$ | 5200 | Ji and Evans (2007) | M | |
| | $1.5\times10^{-1}$ | 4400 | Falabella et al. (2006) | M | 11, 338 |
| | $2.7\times10^{-2}$ | 12000 | Strekowski and George (2005) | M | |
| | $1.7\times10^{-1}$ | | Straver and de Loos (2005) | M | |
| | $1.5\times10^{-1}$ | 4500 | Chai et al. (2005) | M | 11 |
| | | | Cheng et al. (2004) | M | 328 |
| | | | Cheng et al. (2003) | M | 328 |
| | $1.1\times10^{-1}$ | | Karl et al. (2003) | M | |
| | $1.3\times10^{-1}$ | 4300 | Hovorka et al. (2002) | M | 11 |
| | $9.9\times10^{-2}$ | | van Ruth et al. (2002) | M | 14 |
| | $1.0\times10^{-1}$ | | van Ruth and Villeneuve (2002) | M | 14, 361 |
| | $9.0\times10^{-2}$ | | van Ruth et al. (2001) | M | 14 |
| | $1.6\times10^{-1}$ | | Kim et al. (2000) | M | |
| | $1.6\times10^{-1}$ | | Welke et al. (1998) | M | |
| | $1.9\times10^{-1}$ | | Chaintreau et al. (1995) | M | |
| | $1.4\times10^{-1}$ | 4700 | Ettre et al. (1993) | M | 11 |
| | $1.9\times10^{-1}$ | 5000 | Zhou and Mopper (1990) | M | 456 |
| | $6.8\times10^{-2}$ | -5100 | Ashworth et al. (1988) | M | 33, 278 |
| | $1.3\times10^{-1}$ | | Hellmann (1987) | M | 87 |
| | $1.8\times10^{-1}$ | | Park et al. (1987) | M | |
| | $1.7\times10^{-1}$ | 5700 | Snider and Dawson (1985) | M | |
| | $1.4\times10^{-1}$ | | Hawthorne et al. (1985) | M | |
| | $1.2\times10^{-1}$ | | Richon et al. (1985) | M | 38 |
| | $3.2\times10^{-1}$ | | Ioffe et al. (1984) | M | 80 |
| | $1.0\times10^{-1}$ | | Friant and Suffet (1979) | M | 38 |
| | $9.8\times10^{-2}$ | | Sato and Nakajima (1979a) | M | 14 |
| | $1.8\times10^{-1}$ | | Vitenberg et al. (1975) | M | |
| | $1.1\times10^{-1}$ | | Vitenberg et al. (1974) | M | |
| | $1.9\times10^{-1}$ | | Rohrschneider (1973) | M | |
| | $2.1\times10^{-1}$ | | Buttery et al. (1969) | M | |
| | $1.1\times10^{-2}$ | | Abraham and Acree (2007) | V | |
| | $2.8\times10^{-1}$ | | Mackay et al. (2006c) | V | |
| | $2.8\times10^{-1}$ | | Mackay et al. (1995) | V | |
| | $2.6\times10^{-1}$ | | Hwang et al. (1992) | V | |
| | $8.6\times10^{-2}$ | 3700 | Djerki and Laub (1988) | V | |
| | $1.6\times10^{-1}$ | | Rathbun and Tai (1982) | V | |





Table A3.6: Ketones (RCOR) (. . . continued)

| Substance<br>Formula<br>(Trivial Name)<br>[CAS Registry Number]<br>InChIKey | $H_s^{cp}$<br>(at $T^{\ominus}$)<br>$\left[\dfrac{\mathrm{mol}}{\mathrm{m}^3\,\mathrm{Pa}}\right]$ | $\dfrac{\mathrm{d}\ln H_s^{cp}}{\mathrm{d}(1/T)}$<br><br>[K] | Reference | Type | Note |
|---|---|---|---|---|---|
| | $7.1\times10^{-2}$ | | Hine and Weimar (1965) | R | |
| | $2.1\times10^{-1}$ | 5500 | Bagno et al. (1991) | T | 473 |
| | | 5500 | Della Gatta et al. (1981) | T | |
| | $7.6\times10^{-2}$ | | Yaws (2003) | X | 258 |
| | $7.6\times10^{-2}$ | | Yaws (2003) | X | 237 |
| | $7.1\times10^{-2}$ | 5800 | Janini and Quaddora (1986) | X | 298 |
| | $2.3\times10^{-1}$ | | Mackay et al. (1995) | C | |
| | $4.1\times10^{-1}$ | | Harrison et al. (1993) | C | |
| | $1.9\times10^{-1}$ | | Cabani et al. (1981) | C | |
| | $2.3\times10^{-1}$ | | Dupeux et al. (2022) | Q | 259 |
| | $1.8\times10^{-1}$ | | Hayer et al. (2022) | Q | 20 |
| | $1.6\times10^{-1}$ | | Keshavarz et al. (2022) | Q | |
| | $3.2\times10^{-2}$ | | Duchowicz et al. (2020) | Q | |
| | $5.1\times10^{-2}$ | | Wang et al. (2017) | Q | 80, 238 |
| | $2.5\times10^{-1}$ | | Wang et al. (2017) | Q | 80, 239 |
| | $3.6\times10^{-1}$ | | Wang et al. (2017) | Q | 80, 240 |
| | $2.1\times10^{-1}$ | | Li et al. (2014) | Q | 241 |
| | $2.1\times10^{-2}$ | | Gharagheizi et al. (2012) | Q | |
| | $2.0\times10^{-1}$ | | Raventos-Duran et al. (2010) | Q | 242, 243 |
| | $1.6\times10^{-1}$ | | Raventos-Duran et al. (2010) | Q | 244 |
| | $1.6\times10^{-1}$ | | Raventos-Duran et al. (2010) | Q | 245 |
| | $8.4\times10^{-2}$ | | Gharagheizi et al. (2010) | Q | 246 |
| | $1.3\times10^{-1}$ | | Hilal et al. (2008) | Q | |
| | $3.0\times10^{-1}$ | | Modarresi et al. (2007) | Q | 67 |
| | | 5900 | Kühne et al. (2005) | Q | |
| | $1.7\times10^{-1}$ | | Yaffe et al. (2003) | Q | 248, 249 |
| | $1.4\times10^{-1}$ | | English and Carroll (2001) | Q | 230, 231 |
| | $3.0\times10^{-2}$ | | Katritzky et al. (1998) | Q | |
| | $1.6\times10^{-1}$ | | Nirmalakhandan et al. (1997) | Q | |
| | $1.5\times10^{-1}$ | | Suzuki et al. (1992) | Q | 232 |
| | $1.7\times10^{-1}$ | | Duchowicz et al. (2020) | ? | 185, 21 |
| | $1.0\times10^{-1}$ | | Mackay et al. (2006c) | ? | 21 |
| | | 5300 | Kühne et al. (2005) | ? | |
| | $3.3\times10^{-1}$ | | Yaws (1999) | ? | 21 |
| | $1.5\times10^{-1}$ | | Yaws et al. (1998) | ? | |
| | $1.0\times10^{-1}$ | | Abraham and Weathersby (1994) | ? | 21 |
| | $3.1\times10^{-1}$ | | Betterton (1991) | ? | |
| | $2.1\times10^{-1}$ | | Abraham et al. (1990) | ? | |
| butanone-1,1,1,3,3-d5<br>$C_2H_5COCH_3$<br>(methyl ethyl ketone-d5; MEK-d5)<br>[24313-50-6]<br>ZWEHNKRNPOVVGH-PDWRLMEDSA-N | $3.7\times10^{-1}$ | 8200 | Hiatt (2013) | M | |





Table A3.6: Ketones (RCOR) (... continued)

| Substance<br>Formula<br>(Trivial Name)<br>[CAS Registry Number]<br>InChIKey | $H_s^{cp}$<br>(at $T^{\ominus}$)<br>$\left[\dfrac{\text{mol}}{\text{m}^3\,\text{Pa}}\right]$ | $\dfrac{\text{d}\ln H_s^{cp}}{\text{d}(1/T)}$<br><br>[K] | Reference | Type | Note |
|---|---|---|---|---|---|
| 2-pentanone | $1.3\times10^{-1}$ | 6000 | Brockbank (2013) | L | 1 |
| $C_3H_7COCH_3$ | $1.3\times10^{-1}$ | 5900 | Plyasunov and Shock (2001) | L | |
| [107-87-9] | $1.3\times10^{-1}$ | 5900 | Hovorka et al. (2019) | M | |
| XNLICIUVMPYHGG-UHFFFAOYSA-N | $1.6\times10^{-1}$ | 5700 | Ji and Evans (2007) | M | |
| | $1.0\times10^{-1}$ | 4600 | Falabella et al. (2006) | M | 11, 338 |
| | $8.6\times10^{-2}$ | | Straver and de Loos (2005) | M | |
| | $1.0\times10^{-1}$ | 4800 | Chai et al. (2005) | M | 11 |
| | $1.1\times10^{-1}$ | | Kim et al. (2000) | M | |
| | $1.2\times10^{-1}$ | | Shiu and Mackay (1997) | M | |
| | $9.0\times10^{-2}$ | | Hawthorne et al. (1985) | M | |
| | $6.4\times10^{-2}$ | | Sato and Nakajima (1979a) | M | 14 |
| | $1.7\times10^{-1}$ | | Vitenberg et al. (1974) | M | |
| | $1.1\times10^{-1}$ | | Vitenberg et al. (1974) | M | 478 |
| | $1.6\times10^{-1}$ | | Buttery et al. (1969) | M | |
| | $9.2\times10^{-2}$ | | Nelson and Hoff (1968) | M | 297 |
| | $1.5\times10^{-1}$ | | Mackay et al. (2006c) | V | |
| | $5.9\times10^{-2}$ | | Philippe et al. (2003) | V | 14 |
| | $1.5\times10^{-1}$ | | Shiu and Mackay (1997) | V | |
| | $1.5\times10^{-1}$ | | Mackay et al. (1995) | V | |
| | $2.4\times10^{-1}$ | 4300 | Djerki and Laub (1988) | V | |
| | $2.6\times10^{-1}$ | | Rathbun and Tai (1982) | V | |
| | $3.1\times10^{-1}$ | | Amoore and Buttery (1978) | V | |
| | | 5900 | Della Gatta et al. (1981) | T | |
| | $1.4\times10^{-1}$ | | Yaws (2003) | X | 258 |
| | $9.1\times10^{-2}$ | 4600 | Janini and Quaddora (1986) | X | 298 |
| | $1.7\times10^{-1}$ | | Mackay et al. (1995) | C | |
| | $2.5\times10^{-1}$ | | Dupeux et al. (2022) | Q | 259 |
| | $8.7\times10^{-2}$ | | Hayer et al. (2022) | Q | 20 |
| | $2.2\times10^{-1}$ | | Keshavarz et al. (2022) | Q | |
| | $3.2\times10^{-2}$ | | Duchowicz et al. (2020) | Q | 299 |
| | $4.3\times10^{-2}$ | | Wang et al. (2017) | Q | 80, 238 |
| | $1.6\times10^{-1}$ | | Wang et al. (2017) | Q | 80, 239 |
| | $1.8\times10^{-1}$ | | Wang et al. (2017) | Q | 80, 240 |
| | $1.6\times10^{-1}$ | | Raventos-Duran et al. (2010) | Q | 242, 243 |
| | $7.8\times10^{-2}$ | | Raventos-Duran et al. (2010) | Q | 244 |
| | $1.2\times10^{-1}$ | | Raventos-Duran et al. (2010) | Q | 245 |
| | $1.0\times10^{-1}$ | | Hilal et al. (2008) | Q | |
| | $1.7\times10^{-1}$ | | Modarresi et al. (2007) | Q | 67 |
| | | 6200 | Kühne et al. (2005) | Q | |
| | $1.4\times10^{-1}$ | | Yaffe et al. (2003) | Q | 248, 272 |
| | $1.9\times10^{-2}$ | | Katritzky et al. (1998) | Q | |
| | $1.2\times10^{-1}$ | | Nirmalakhandan et al. (1997) | Q | |
| | $1.2\times10^{-1}$ | | Russell et al. (1992) | Q | 279 |
| | $1.2\times10^{-1}$ | | Suzuki et al. (1992) | Q | 232 |
| | $1.2\times10^{-1}$ | | Duchowicz et al. (2020) | ? | 185, 21 |
| | $1.6\times10^{-1}$ | | Mackay et al. (2006c) | ? | 21 |
| | | 6500 | Kühne et al. (2005) | ? | |



Table A3.6: Ketones (RCOR) (... continued)

| Substance Formula (Trivial Name) [CAS Registry Number] InChIKey | $H_s^{cp}$ (at $T^\ominus$) $\left[\dfrac{\mathrm{mol}}{\mathrm{m^3\,Pa}}\right]$ | $\dfrac{\mathrm{d}\ln H_s^{cp}}{\mathrm{d}(1/T)}$ [K] | Reference | Type | Note |
|---|---|---|---|---|---|
| | $1.4\times10^{-1}$ | | Yaws (1999) | ? | 21 |
| | $1.3\times10^{-1}$ | | Yaws et al. (1998) | ? | |
| | $6.7\times10^{-2}$ | | Abraham and Weathersby (1994) | ? | 21 |
| | $1.5\times10^{-1}$ | | Abraham et al. (1990) | ? | |
| | $3.1\times10^{-1}$ | | Mackay and Yeun (1983) | ? | |
| 3-pentanone | $8.9\times10^{-2}$ | 5800 | Brockbank (2013) | L | 1, 479 |
| $C_2H_5COC_2H_5$ | $1.1\times10^{-1}$ | 6000 | Plyasunov and Shock (2001) | L | |
| (diethyl ketone) | $1.1\times10^{-1}$ | 5900 | Hovorka et al. (2019) | M | |
| [96-22-0] | $2.1\times10^{-1}$ | | O'Farrell and Waghorne (2010) | M | |
| FDPIMTJIUBPUKL-UHFFFAOYSA-N | $1.6\times10^{-1}$ | 5600 | Ji and Evans (2007) | M | |
| | $7.0\times10^{-2}$ | | Sato and Nakajima (1979a) | M | 14 |
| | $9.7\times10^{-5}$ | | Saylor et al. (1938) | M | 38 |
| | $8.4\times10^{-2}$ | | Mackay et al. (2006c) | V | |
| | $1.2\times10^{-1}$ | | Mackay et al. (1995) | V | |
| | $8.4\times10^{-2}$ | | Mackay et al. (1995) | V | |
| | $2.8\times10^{-1}$ | | Rathbun and Tai (1982) | V | |
| | $1.3\times10^{-1}$ | 6000 | Bagno et al. (1991) | T | 473 |
| | | 6000 | Della Gatta et al. (1981) | T | |
| | $8.2\times10^{-2}$ | | Yaws (2003) | X | 258 |
| | $8.2\times10^{-2}$ | | Yaws (2003) | X | 237 |
| | $2.0\times10^{-1}$ | 9200 | Janini and Quaddora (1986) | X | 298 |
| | $1.1\times10^{-1}$ | | Howard (1993) | X | 412 |
| | $1.3\times10^{-1}$ | | Cabani et al. (1981) | C | |
| | $8.6\times10^{-2}$ | | Dupeux et al. (2022) | Q | 259 |
| | $2.2\times10^{-1}$ | | Keshavarz et al. (2022) | Q | |
| | $3.2\times10^{-2}$ | | Duchowicz et al. (2020) | Q | |
| | $4.3\times10^{-2}$ | | Wang et al. (2017) | Q | 80, 238 |
| | $1.3\times10^{-1}$ | | Wang et al. (2017) | Q | 80, 239 |
| | $6.6\times10^{-2}$ | | Wang et al. (2017) | Q | 80, 240 |
| | $1.6\times10^{-1}$ | | Raventos-Duran et al. (2010) | Q | 242, 243 |
| | $7.8\times10^{-2}$ | | Raventos-Duran et al. (2010) | Q | 244 |
| | $1.2\times10^{-1}$ | | Raventos-Duran et al. (2010) | Q | 245 |
| | $7.5\times10^{-2}$ | | Gharagheizi et al. (2010) | Q | 246 |
| | $9.2\times10^{-2}$ | | Hilal et al. (2008) | Q | |
| | $1.9\times10^{-1}$ | | Modarresi et al. (2007) | Q | 67 |
| | | 6200 | Kühne et al. (2005) | Q | |
| | $1.4\times10^{-1}$ | | Yaffe et al. (2003) | Q | 248, 249 |
| | $4.3\times10^{-2}$ | | Yao et al. (2002) | Q | 229 |
| | $1.1\times10^{-1}$ | | English and Carroll (2001) | Q | 230, 274 |
| | $2.7\times10^{-2}$ | | Katritzky et al. (1998) | Q | |
| | $1.2\times10^{-1}$ | | Nirmalakhandan et al. (1997) | Q | |
| | $2.0\times10^{-1}$ | | Duchowicz et al. (2020) | ? | 185, 21 |
| | | 6800 | Kühne et al. (2005) | ? | |
| | $8.2\times10^{-2}$ | | Yaws (1999) | ? | 21 |
| | $1.2\times10^{-1}$ | | Yaws et al. (1998) | ? | |
| | $7.3\times10^{-2}$ | | Abraham and Weathersby (1994) | ? | 21 |
| | $1.3\times10^{-1}$ | | Abraham et al. (1990) | ? | |





Table A3.6: Ketones (RCOR) (...continued)

| Substance<br>Formula<br>(Trivial Name)<br>[CAS Registry Number]<br>InChIKey | $H_s^{cp}$<br>(at $T^{\ominus}$)<br>$\left[\dfrac{\text{mol}}{\text{m}^3\,\text{Pa}}\right]$ | $\dfrac{\text{d}\ln H_s^{cp}}{\text{d}(1/T)}$<br><br>[K] | Reference | Type | Note |
|---|---|---|---|---|---|
| 1-cyclopropyl-ethanone | $9.5\times10^{-1}$ | 5900 | Bagno et al. (1991) | T | 473 |
| $C_5H_8O$ | | 5900 | Della Gatta et al. (1981) | T | |
| (cyclopropyl methyl ketone) | $2.2\times10^{-1}$ | | Keshavarz et al. (2022) | Q | |
| [765-43-5] | $9.9\times10^{-2}$ | | Duchowicz et al. (2020) | Q | 299 |
| HVCFCNAITDHQFX-UHFFFAOYSA-N | $4.8\times10^{-1}$ | | Hilal et al. (2008) | Q | |
| | $5.2\times10^{-1}$ | | Modarresi et al. (2007) | Q | 67 |
| | $3.1\times10^{-1}$ | | English and Carroll (2001) | Q | 230, 274 |
| | $6.4\times10^{-1}$ | | Nirmalakhandan et al. (1997) | Q | |
| | $9.7\times10^{-1}$ | | Duchowicz et al. (2020) | ? | 185, 21 |
| 3-methyl-2-butanone | $1.1\times10^{-1}$ | 5800 | Brockbank (2013) | L | 1 |
| $C_5H_{10}O$ | $9.1\times10^{-2}$ | 5700 | Plyasunov and Shock (2001) | L | |
| (isopropyl methyl ketone) | $9.3\times10^{-2}$ | 5600 | Hovorka et al. (2019) | M | |
| [563-80-4] | $1.0\times10^{-1}$ | | Duchowicz et al. (2020) | V | 186 |
| SYBYTAAJFKOIEJ-UHFFFAOYSA-N | $8.7\times10^{-2}$ | | HSDB (2015) | V | |
| | $9.6\times10^{-2}$ | | Cabani et al. (1981) | V | |
| | $9.0\times10^{-2}$ | 5700 | Bagno et al. (1991) | T | 473 |
| | | 5700 | Della Gatta et al. (1981) | T | |
| | $1.3\times10^{-2}$ | | Duchowicz et al. (2020) | Q | |
| | $4.9\times10^{-2}$ | | Wang et al. (2017) | Q | 80, 238 |
| | $1.3\times10^{-1}$ | | Wang et al. (2017) | Q | 80, 239 |
| | $2.5\times10^{-1}$ | | Wang et al. (2017) | Q | 80, 240 |
| | $1.6\times10^{-1}$ | | Raventos-Duran et al. (2010) | Q | 271, 243 |
| | $6.2\times10^{-2}$ | | Raventos-Duran et al. (2010) | Q | 244 |
| | $1.2\times10^{-1}$ | | Raventos-Duran et al. (2010) | Q | 245 |
| | $8.4\times10^{-2}$ | | Hilal et al. (2008) | Q | |
| | $2.1\times10^{-1}$ | | Modarresi et al. (2007) | Q | 67 |
| | | 5300 | Kühne et al. (2005) | Q | |
| | $9.7\times10^{-2}$ | | Yaffe et al. (2003) | Q | 248, 249 |
| | $6.4\times10^{-2}$ | | English and Carroll (2001) | Q | 230, 231 |
| | $2.5\times10^{-2}$ | | Katritzky et al. (1998) | Q | |
| | $1.0\times10^{-1}$ | | Nirmalakhandan et al. (1997) | Q | |
| | | 7200 | Kühne et al. (2005) | ? | |
| | $1.1\times10^{-1}$ | | Yaws (1999) | ? | 21 |
| | $1.1\times10^{-1}$ | | Yaws et al. (1998) | ? | |
| | $9.7\times10^{-2}$ | | Abraham et al. (1990) | ? | |
| cyclopentanone | 1.2 | 5800 | Brockbank (2013) | L | 1 |
| $C_5H_8O$ | 2.8 | | O'Farrell and Waghorne (2010) | M | |
| [120-92-3] | $8.2\times10^{-1}$ | 4900 | Hovorka et al. (2002) | M | 11 |
| BGTOWKSIORTVQH-UHFFFAOYSA-N | $8.2\times10^{-1}$ | | Hawthorne et al. (1985) | M | |
| | $7.4\times10^{-1}$ | | Keshavarz et al. (2022) | Q | |
| | $2.5\times10^{-1}$ | | Duchowicz et al. (2020) | Q | 299 |
| | $6.2\times10^{-1}$ | | Raventos-Duran et al. (2010) | Q | 242, 243 |
| | $9.9\times10^{-1}$ | | Raventos-Duran et al. (2010) | Q | 244 |
| | $2.5\times10^{-1}$ | | Raventos-Duran et al. (2010) | Q | 245 |
| | 1.1 | | Hilal et al. (2008) | Q | |
| | $4.7\times10^{-1}$ | | Modarresi et al. (2007) | Q | 67 |




Table A3.6: Ketones (RCOR) (...continued)

| Substance Formula (Trivial Name) [CAS Registry Number] InChIKey | $H_s^{cp}$ (at $T^{\ominus}$) $\left[\dfrac{\mathrm{mol}}{\mathrm{m^3\,Pa}}\right]$ | $\dfrac{\mathrm{d}\ln H_s^{cp}}{\mathrm{d}(1/T)}$ [K] | Reference | Type | Note |
|---|---|---|---|---|---|
| | | 5800 | Kühne et al. (2005) | Q | |
| | 1.0 | | Yaffe et al. (2003) | Q | 248, 249 |
| | $7.5\times10^{-1}$ | | English and Carroll (2001) | Q | 230, 260 |
| | $6.9\times10^{-2}$ | | Katritzky et al. (1998) | Q | |
| | $7.2\times10^{-1}$ | | Nirmalakhandan et al. (1997) | Q | |
| | $9.9\times10^{-1}$ | | Duchowicz et al. (2020) | ? | 185, 21 |
| | | 5600 | Kühne et al. (2005) | ? | |
| | 1.1 | | Abraham et al. (1990) | ? | |
| 2-hexanone $C_6H_{12}O$ [591-78-6] QQZOPKMRPOGIEB-UHFFFAOYSA-N | $1.1\times10^{-1}$ | 6500 | Brockbank (2013) | L | 1 |
| | $9.8\times10^{-2}$ | 6300 | Plyasunov and Shock (2001) | L | |
| | $9.9\times10^{-2}$ | 6300 | Hovorka et al. (2019) | M | |
| | $1.5\times10^{-1}$ | 8600 | Hiatt (2013) | M | |
| | $7.9\times10^{-2}$ | 4800 | Falabella et al. (2006) | M | 11, 338 |
| | $1.1\times10^{-1}$ | | Straver and de Loos (2005) | M | |
| | $8.6\times10^{-2}$ | 5100 | Chai et al. (2005) | M | 11 |
| | $4.3\times10^{-2}$ | | Sato and Nakajima (1979a) | M | 14 |
| | $1.1\times10^{-1}$ | | Duchowicz et al. (2020) | V | 186 |
| | $1.1\times10^{-1}$ | | HSDB (2015) | V | |
| | $1.1\times10^{-1}$ | | Mackay et al. (2006c) | V | |
| | $1.1\times10^{-1}$ | | Mackay et al. (1995) | V | |
| | $1.0\times10^{-1}$ | | Meylan and Howard (1991) | V | |
| | $6.5\times10^{-1}$ | 4900 | Djerki and Laub (1988) | V | |
| | $1.0\times10^{-1}$ | | Cabani et al. (1981) | V | |
| | | 6200 | Della Gatta et al. (1981) | T | |
| | $1.2\times10^{-1}$ | | Yaws (2003) | X | 258 |
| | $1.0\times10^{-1}$ | | Howard (1993) | X | 412 |
| | $1.7\times10^{-1}$ | | Dupeux et al. (2022) | Q | 259 |
| | $3.2\times10^{-2}$ | | Duchowicz et al. (2020) | Q | |
| | $3.8\times10^{-2}$ | | Wang et al. (2017) | Q | 80, 238 |
| | $1.1\times10^{-1}$ | | Wang et al. (2017) | Q | 80, 239 |
| | $1.7\times10^{-1}$ | | Wang et al. (2017) | Q | 80, 240 |
| | $8.2\times10^{-2}$ | | Hilal et al. (2008) | Q | |
| | $1.5\times10^{-1}$ | | Modarresi et al. (2007) | Q | 67 |
| | | 6600 | Kühne et al. (2005) | Q | |
| | $7.9\times10^{-2}$ | | Yaffe et al. (2003) | Q | 248, 272 |
| | $8.4\times10^{-2}$ | | English and Carroll (2001) | Q | 230, 231 |
| | $2.4\times10^{-2}$ | | Katritzky et al. (1998) | Q | |
| | $9.2\times10^{-2}$ | | Nirmalakhandan et al. (1997) | Q | |
| | $8.5\times10^{-2}$ | | Meylan and Howard (1991) | Q | |
| | | 6200 | Kühne et al. (2005) | ? | |
| | $1.2\times10^{-1}$ | | Yaws (1999) | ? | 21 |
| | $1.2\times10^{-1}$ | | Yaws et al. (1998) | ? | |
| | $4.5\times10^{-2}$ | | Abraham and Weathersby (1994) | ? | 21 |
| | $1.0\times10^{-1}$ | | Abraham et al. (1990) | ? | |





Table A3.6: Ketones (RCOR) (. . . continued)

| Substance Formula (Trivial Name) [CAS Registry Number] InChIKey | $H_s^{cp}$ (at $T^{\ominus}$) $\left[\dfrac{\text{mol}}{\text{m}^3\,\text{Pa}}\right]$ | $\dfrac{\text{d}\ln H_s^{cp}}{\text{d}(1/T)}$ [K] | Reference | Type | Note |
|---|---|---|---|---|---|
| 2-hexanone-1,1,1,3,3-d5 $C_6H_{12}O$ [4840-82-8] QQZOPKMRPOGIEB-ZTIZGVCASA-N | $1.7\times10^{-1}$ | 9000 | Hiatt (2013) | M | |
| 3-hexanone $C_6H_{12}O$ [589-38-8] PFCHFHIRKBAQGU-UHFFFAOYSA-N | $7.9\times10^{-2}$ | 7100 | Brockbank (2013) | L | 1 |
| | $7.5\times10^{-2}$ | | Plyasunov and Shock (2001) | L | |
| | $7.6\times10^{-2}$ | 6400 | Hovorka et al. (2019) | M | |
| | | | Dewulf et al. (1999) | M | 362 |
| | $7.9\times10^{-2}$ | | Duchowicz et al. (2020) | V | 186 |
| | $8.0\times10^{-2}$ | | Yaws (2003) | X | 237 |
| | $3.2\times10^{-2}$ | | Duchowicz et al. (2020) | Q | |
| | $3.8\times10^{-2}$ | | Wang et al. (2017) | Q | 80, 238 |
| | $8.9\times10^{-2}$ | | Wang et al. (2017) | Q | 80, 239 |
| | $1.5\times10^{-1}$ | | Wang et al. (2017) | Q | 80, 240 |
| | $6.2\times10^{-2}$ | | Gharagheizi et al. (2010) | Q | 246 |
| | $6.9\times10^{-2}$ | | Hilal et al. (2008) | Q | |
| | $1.6\times10^{-1}$ | | Modarresi et al. (2007) | Q | 67 |
| | | 6600 | Kühne et al. (2005) | Q | |
| | $7.9\times10^{-2}$ | | Yaffe et al. (2003) | Q | 248, 249 |
| | $2.1\times10^{-2}$ | | Katritzky et al. (1998) | Q | |
| | | 5800 | Kühne et al. (2005) | ? | |
| | $8.0\times10^{-2}$ | | Yaws (1999) | ? | 21 |
| | $8.0\times10^{-2}$ | | Yaws et al. (1998) | ? | |
| 3-methyl-2-pentanone $C_6H_{12}O$ [565-61-7] UIHCLUNTQKBZGK-UHFFFAOYSA-N | $8.4\times10^{-2}$ | | Plyasunov and Shock (2001) | L | |
| | $7.9\times10^{-2}$ | 6000 | Hovorka et al. (2019) | M | |
| | $1.3\times10^{-1}$ | | Duchowicz et al. (2020) | V | 186 |
| | $1.3\times10^{-2}$ | | Duchowicz et al. (2020) | Q | |
| | $4.0\times10^{-2}$ | | Wang et al. (2017) | Q | 80, 238 |
| | $1.1\times10^{-1}$ | | Wang et al. (2017) | Q | 80, 239 |
| | $1.8\times10^{-1}$ | | Wang et al. (2017) | Q | 80, 240 |
| | $9.9\times10^{-2}$ | | Raventos-Duran et al. (2010) | Q | 242, 243 |
| | $6.2\times10^{-2}$ | | Raventos-Duran et al. (2010) | Q | 244 |
| | $7.8\times10^{-2}$ | | Raventos-Duran et al. (2010) | Q | 245 |
| | $7.3\times10^{-2}$ | | Hilal et al. (2008) | Q | |
| | $2.2\times10^{-1}$ | | Modarresi et al. (2007) | Q | 67 |
| | $9.6\times10^{-2}$ | | Yaws et al. (1998) | ? | |
| 4-methyl-2-pentanone $(CH_3)_2CHCH_2COCH_3$ (methyl isobutyl ketone; MIBK) [108-10-1] NTIZESTWPVYFNL-UHFFFAOYSA-N | $6.9\times10^{-2}$ | 6200 | Brockbank (2013) | L | 1 |
| | $6.9\times10^{-2}$ | 6000 | Plyasunov and Shock (2001) | L | |
| | $6.5\times10^{-2}$ | 6000 | Hovorka et al. (2019) | M | |
| | $3.9\times10^{-2}$ | | Kim and Kim (2014) | M | |
| | $1.0\times10^{-1}$ | 8700 | Hiatt (2013) | M | |
| | $3.9\times10^{-2}$ | | Kim et al. (2000) | M | |
| | $4.7\times10^{-2}$ | | Welke et al. (1998) | M | |
| | $4.3\times10^{-2}$ | 4600 | Kolb et al. (1992) | M | 277 |
| | $2.2\times10^{-2}$ | 110 | Ashworth et al. (1988) | M | 42, 278 |
| | $6.5\times10^{-2}$ | | Hellmann (1987) | M | 87 |





Table A3.6: Ketones (RCOR) (... continued)

| Substance Formula (Trivial Name) [CAS Registry Number] InChIKey | $H_s^{cp}$ (at $T^{\ominus}$) $\left[\dfrac{\mathrm{mol}}{\mathrm{m^3\,Pa}}\right]$ | $\dfrac{\mathrm{d}\ln H_s^{cp}}{\mathrm{d}(1/T)}$ [K] | Reference | Type | Note |
|---|---|---|---|---|---|
| | $3.1\times10^{-2}$ | | Sato and Nakajima (1979a) | M | 14 |
| | $7.2\times10^{-2}$ | | Duchowicz et al. (2020) | V | 186 |
| | $7.0\times10^{-2}$ | | HSDB (2015) | V | |
| | $6.5\times10^{-2}$ | | Mackay et al. (2006c) | V | |
| | $6.5\times10^{-2}$ | | Mackay et al. (1995) | V | |
| | $7.2\times10^{-2}$ | | Hwang et al. (1992) | V | |
| | $1.4\times10^{-1}$ | | Rathbun and Tai (1982) | V | |
| | $7.1\times10^{-2}$ | | Cabani et al. (1981) | V | |
| | $7.3\times10^{-2}$ | | Yaws (2003) | X | 258 |
| | $1.1\times10^{-1}$ | | Howard (1990) | X | 412 |
| | $1.4\times10^{-1}$ | | Dupeux et al. (2022) | Q | 259 |
| | $1.3\times10^{-2}$ | | Duchowicz et al. (2020) | Q | |
| | $4.0\times10^{-2}$ | | Wang et al. (2017) | Q | 80, 238 |
| | $1.3\times10^{-1}$ | | Wang et al. (2017) | Q | 80, 239 |
| | $2.1\times10^{-1}$ | | Wang et al. (2017) | Q | 80, 240 |
| | $8.8\times10^{-2}$ | | Hilal et al. (2008) | Q | |
| | $1.4\times10^{-1}$ | | Modarresi et al. (2007) | Q | 67 |
| | | 6600 | Kühne et al. (2005) | Q | |
| | $7.3\times10^{-2}$ | | Yaffe et al. (2003) | Q | 248, 249 |
| | $5.1\times10^{-2}$ | | English and Carroll (2001) | Q | 230, 260 |
| | $1.7\times10^{-2}$ | | Katritzky et al. (1998) | Q | |
| | $7.9\times10^{-2}$ | | Nirmalakhandan et al. (1997) | Q | |
| | | 5700 | Kühne et al. (2005) | ? | |
| | $7.3\times10^{-2}$ | | Yaws (1999) | ? | 21 |
| | $7.2\times10^{-2}$ | | Yaws et al. (1998) | ? | |
| | $3.2\times10^{-2}$ | | Abraham and Weathersby (1994) | ? | 21 |
| | $3.0\times10^{-1}$ | | Betterton (1991) | ? | |
| | $7.0\times10^{-2}$ | | Abraham et al. (1990) | ? | |
| 2-methyl-3-pentanone $C_6H_{12}O$ [565-69-5] HYTRYEXINDDXJK-UHFFFAOYSA-N | $5.3\times10^{-2}$ | | Plyasunov and Shock (2001) | L | |
| | $5.2\times10^{-2}$ | 6100 | Hovorka et al. (2019) | M | |
| | $6.4\times10^{-2}$ | | Duchowicz et al. (2020) | V | 186 |
| | $1.3\times10^{-2}$ | | Duchowicz et al. (2020) | Q | |
| | $4.0\times10^{-2}$ | | Wang et al. (2017) | Q | 80, 238 |
| | $9.3\times10^{-2}$ | | Wang et al. (2017) | Q | 80, 239 |
| | $1.1\times10^{-1}$ | | Wang et al. (2017) | Q | 80, 240 |
| | $9.9\times10^{-2}$ | | Raventos-Duran et al. (2010) | Q | 242, 243 |
| | $4.9\times10^{-2}$ | | Raventos-Duran et al. (2010) | Q | 244 |
| | $7.8\times10^{-2}$ | | Raventos-Duran et al. (2010) | Q | 245 |
| | $6.5\times10^{-2}$ | | Hilal et al. (2008) | Q | |
| | $1.6\times10^{-1}$ | | Modarresi et al. (2007) | Q | 67 |
| | $9.7\times10^{-2}$ | | Yaffe et al. (2003) | Q | 248, 272 |
| | $2.3\times10^{-2}$ | | Katritzky et al. (1998) | Q | |
| | $6.4\times10^{-2}$ | | Yaws et al. (1998) | ? | |





Table A3.6: Ketones (RCOR) (... continued)

| Substance<br>Formula<br>(Trivial Name)<br>[CAS Registry Number]<br>InChIKey | $H_s^{cp}$<br>(at $T^{\ominus}$)<br>$\left[\dfrac{\text{mol}}{\text{m}^3\,\text{Pa}}\right]$ | $\dfrac{\text{d}\ln H_s^{cp}}{\text{d}(1/T)}$<br><br>[K] | Reference | Type | Note |
|---|---|---|---|---|---|
| 3,3-dimethyl-2-butanone | $4.3\times10^{-2}$ | 5800 | Brockbank (2013) | L | 1 |
| $C_6H_{12}O$ | $4.5\times10^{-2}$ | 5700 | Plyasunov and Shock (2001) | L | |
| (*tert*-butyl methyl ketone) | $4.3\times10^{-2}$ | 5700 | Hovorka et al. (2019) | M | |
| [75-97-8] | $4.5\times10^{-2}$ | | Duchowicz et al. (2020) | V | 186 |
| PJGSXYOJTGTZAV-UHFFFAOYSA-N | $4.5\times10^{-2}$ | | HSDB (2015) | V | |
| | $7.6\times10^{-2}$ | 6000 | Bagno et al. (1991) | T | 473 |
| | | 6000 | Della Gatta et al. (1981) | T | |
| | $5.5\times10^{-3}$ | | Duchowicz et al. (2020) | Q | |
| | $2.8\times10^{-2}$ | | Wang et al. (2017) | Q | 80, 238 |
| | $7.1\times10^{-2}$ | | Wang et al. (2017) | Q | 80, 239 |
| | $1.8\times10^{-1}$ | | Wang et al. (2017) | Q | 80, 240 |
| | $9.9\times10^{-2}$ | | Raventos-Duran et al. (2010) | Q | 242, 243 |
| | $3.1\times10^{-2}$ | | Raventos-Duran et al. (2010) | Q | 244 |
| | $7.8\times10^{-2}$ | | Raventos-Duran et al. (2010) | Q | 245 |
| | $4.7\times10^{-2}$ | | Hilal et al. (2008) | Q | |
| | $2.8\times10^{-1}$ | | Modarresi et al. (2007) | Q | 67 |
| | | 5700 | Kühne et al. (2005) | Q | |
| | $7.9\times10^{-2}$ | | Yaffe et al. (2003) | Q | 248, 249 |
| | $4.5\times10^{-2}$ | | English and Carroll (2001) | Q | 230, 231 |
| | $2.7\times10^{-2}$ | | Katritzky et al. (1998) | Q | |
| | $7.9\times10^{-2}$ | | Nirmalakhandan et al. (1997) | Q | |
| | | 5400 | Kühne et al. (2005) | ? | |
| | $6.4\times10^{-2}$ | | Yaws et al. (1998) | ? | |
| cyclohexanone | 1.5 | 6400 | Brockbank (2013) | L | 1 |
| $C_6H_{10}O$ | $8.6\times10^{-1}$ | 5100 | Hovorka et al. (2002) | M | 11 |
| [108-94-1] | $8.2\times10^{-1}$ | | Hawthorne et al. (1985) | M | |
| JHIVVAPYMSGYDF-UHFFFAOYSA-N | 1.1 | | HSDB (2015) | V | |
| | $3.8\times10^{-1}$ | | Mackay et al. (2006c) | V | |
| | $3.8\times10^{-1}$ | | Mackay et al. (1995) | V | |
| | $4.4\times10^{-1}$ | | Meylan and Howard (1991) | V | |
| | 2.2 | | Yaws (2003) | X | 258 |
| | 2.6 | | Dupeux et al. (2022) | Q | 259 |
| | 1.0 | | Keshavarz et al. (2022) | Q | |
| | $2.5\times10^{-1}$ | | Duchowicz et al. (2020) | Q | 184 |
| | $1.0\times10^{-1}$ | | Wang et al. (2017) | Q | 80, 238 |
| | 2.0 | | Wang et al. (2017) | Q | 80, 239 |
| | 5.9 | | Wang et al. (2017) | Q | 80, 240 |
| | $3.9\times10^{-1}$ | | Raventos-Duran et al. (2010) | Q | 242, 243 |
| | $9.9\times10^{-1}$ | | Raventos-Duran et al. (2010) | Q | 244 |
| | $2.0\times10^{-1}$ | | Raventos-Duran et al. (2010) | Q | 245 |
| | 1.0 | | Hilal et al. (2008) | Q | |
| | 1.1 | | Modarresi et al. (2007) | Q | 67 |
| | | 6200 | Kühne et al. (2005) | Q | |
| | 1.2 | | Yaffe et al. (2003) | Q | 248, 249 |
| | $9.0\times10^{-1}$ | | English and Carroll (2001) | Q | 230, 231 |
| | $6.2\times10^{-2}$ | | Katritzky et al. (1998) | Q | |
| | $5.6\times10^{-1}$ | | Nirmalakhandan et al. (1997) | Q | |





Table A3.6: Ketones (RCOR) (...continued)

| Substance<br>Formula<br>(Trivial Name)<br>[CAS Registry Number]<br>InChIKey | $H_s^{cp}$ (at $T^{\ominus}$) $\left[\dfrac{\text{mol}}{\text{m}^3\,\text{Pa}}\right]$ | $\dfrac{\text{d}\ln H_s^{cp}}{\text{d}(1/T)}$ [K] | Reference | Type | Note |
|---|---|---|---|---|---|
| | $1.9\times10^{-1}$ | | Meylan and Howard (1991) | Q | |
| | $1.1$ | | Duchowicz et al. (2020) | ? | 185, 21 |
| | | 6300 | Kühne et al. (2005) | ? | |
| | $1.8$ | | Yaws (1999) | ? | 21 |
| | $1.6$ | | Abraham et al. (1990) | ? | |
| 2-heptanone | $6.8\times10^{-2}$ | 6800 | Brockbank (2013) | L | 1 |
| $C_7H_{14}O$ | $7.2\times10^{-2}$ | 6800 | Plyasunov and Shock (2001) | L | |
| [110-43-0] | $5.9\times10^{-2}$ | 5300 | Falabella et al. (2006) | M | 11, 338 |
| CATSNJVOTSVZJV-UHFFFAOYSA-N | $6.8\times10^{-2}$ | 5700 | Chai et al. (2005) | M | 11 |
| | $3.9\times10^{-2}$ | | van Ruth et al. (2002) | M | 14 |
| | $4.1\times10^{-2}$ | | van Ruth and Villeneuve (2002) | M | 14, 361 |
| | $2.5\times10^{-2}$ | | van Ruth et al. (2001) | M | 14 |
| | $6.2\times10^{-2}$ | | Kim et al. (2000) | M | |
| | $5.8\times10^{-2}$ | | Shiu and Mackay (1997) | M | |
| | $3.7\times10^{-2}$ | | Sato and Nakajima (1979a) | M | 14 |
| | $6.8\times10^{-2}$ | | Buttery et al. (1969) | M | |
| | $7.5\times10^{-2}$ | | Mackay et al. (2006c) | V | |
| | $7.5\times10^{-2}$ | | Shiu and Mackay (1997) | V | |
| | $7.5\times10^{-2}$ | | Mackay et al. (1995) | V | |
| | $1.8$ | 5600 | Djerki and Laub (1988) | V | |
| | $1.7\times10^{-1}$ | | Rathbun and Tai (1982) | V | |
| | $7.5\times10^{-2}$ | | Yaws (2003) | X | 258 |
| | $7.4\times10^{-2}$ | | Yaws (2003) | X | 237 |
| | $3.5\times10^{-1}$ | 4500 | Janini and Quaddora (1986) | X | 298 |
| | $1.5\times10^{-1}$ | | Dupeux et al. (2022) | Q | 259 |
| | $5.3\times10^{-2}$ | | Keshavarz et al. (2022) | Q | |
| | $3.2\times10^{-2}$ | | Duchowicz et al. (2020) | Q | 184 |
| | $7.8\times10^{-2}$ | | Raventos-Duran et al. (2010) | Q | 271, 243 |
| | $3.9\times10^{-2}$ | | Raventos-Duran et al. (2010) | Q | 244 |
| | $6.2\times10^{-2}$ | | Raventos-Duran et al. (2010) | Q | 245 |
| | $5.1\times10^{-2}$ | | Gharagheizi et al. (2010) | Q | 246 |
| | $6.2\times10^{-2}$ | | Hilal et al. (2008) | Q | |
| | $1.3\times10^{-1}$ | | Modarresi et al. (2007) | Q | 67 |
| | | 6900 | Kühne et al. (2005) | Q | |
| | $1.2\times10^{-1}$ | | Yaffe et al. (2003) | Q | 248, 272 |
| | $6.5\times10^{-2}$ | | English and Carroll (2001) | Q | 230, 274 |
| | $2.3\times10^{-2}$ | | Katritzky et al. (1998) | Q | |
| | $7.2\times10^{-2}$ | | Nirmalakhandan et al. (1997) | Q | |
| | $6.2\times10^{-2}$ | | Russell et al. (1992) | Q | 279 |
| | $6.9\times10^{-2}$ | | Suzuki et al. (1992) | Q | 232 |
| | $5.8\times10^{-2}$ | | Duchowicz et al. (2020) | ? | 185, 21 |
| | $6.8\times10^{-2}$ | | Mackay et al. (2006c) | ? | 21 |
| | | 6900 | Kühne et al. (2005) | ? | |
| | $7.5\times10^{-2}$ | | Yaws (1999) | ? | 21 |
| | $7.5\times10^{-2}$ | | Yaws et al. (1998) | ? | |
| | $3.9\times10^{-2}$ | | Abraham and Weathersby (1994) | ? | 21 |
| | $6.9\times10^{-2}$ | | Abraham et al. (1990) | ? | |





Table A3.6: Ketones (RCOR) (...continued)

| Substance<br>Formula<br>(Trivial Name)<br>[CAS Registry Number]<br>InChIKey | $H_s^{cp}$<br>(at $T^{\ominus}$)<br>$\left[\dfrac{\mathrm{mol}}{\mathrm{m^3\,Pa}}\right]$ | $\dfrac{\mathrm{d}\ln H_s^{cp}}{\mathrm{d}(1/T)}$<br><br>[K] | Reference | Type | Note |
|---|---|---|---|---|---|
| | $1.1\times10^{-1}$ | | Mackay and Yeun (1983) | ? | |
| 3-heptanone | $7.2\times10^{-2}$ | 7000 | Brockbank (2013) | L | 1 |
| $C_7H_{14}O$ | $1.1\times10^{-1}$ | | HSDB (2015) | V | |
| [106-35-4] | $4.9\times10^{-2}$ | | Yaws (2003) | X | 258 |
| NGAZZOYFWWSOGK-UHFFFAOYSA-N | $5.0\times10^{-2}$ | | Yaws (2003) | X | 237 |
| | $7.3\times10^{-2}$ | | Dupeux et al. (2022) | Q | 259 |
| | $3.1\times10^{-2}$ | | Wang et al. (2017) | Q | 80, 238 |
| | $6.0\times10^{-2}$ | | Wang et al. (2017) | Q | 80, 239 |
| | $1.4\times10^{-1}$ | | Wang et al. (2017) | Q | 80, 240 |
| | $7.8\times10^{-2}$ | | Raventos-Duran et al. (2010) | Q | 242, 243 |
| | $3.1\times10^{-2}$ | | Raventos-Duran et al. (2010) | Q | 244 |
| | $6.2\times10^{-2}$ | | Raventos-Duran et al. (2010) | Q | 245 |
| | $5.1\times10^{-2}$ | | Gharagheizi et al. (2010) | Q | 246 |
| | $1.3\times10^{-1}$ | | Modarresi et al. (2007) | Q | 67 |
| | | 6900 | Kühne et al. (2005) | Q | |
| | $1.2\times10^{-1}$ | | Yaffe et al. (2003) | Q | 248, 249 |
| | $1.4\times10^{-2}$ | | Katritzky et al. (1998) | Q | |
| | | 6000 | Kühne et al. (2005) | ? | |
| | $2.4\times10^{-2}$ | | Yaws et al. (1998) | ? | |
| 4-heptanone | $5.1\times10^{-2}$ | 6800 | Brockbank (2013) | L | 1, 480 |
| $C_7H_{14}O$ | $6.1\times10^{-2}$ | 7000 | Plyasunov and Shock (2001) | L | |
| [123-19-3] | $1.8\times10^{-1}$ | | Duchowicz et al. (2020) | V | 186 |
| HCFAJYNVAYBARA-UHFFFAOYSA-N | $4.1\times10^{-2}$ | | HSDB (2015) | V | |
| | $5.6\times10^{-2}$ | | Cabani et al. (1981) | V | |
| | $3.3\times10^{-2}$ | | Yaws (2003) | X | 258 |
| | $3.3\times10^{-2}$ | | Yaws (2003) | X | 237 |
| | $3.8\times10^{-2}$ | | Dupeux et al. (2022) | Q | 259 |
| | $3.2\times10^{-2}$ | | Duchowicz et al. (2020) | Q | |
| | $7.8\times10^{-2}$ | | Raventos-Duran et al. (2010) | Q | 242, 243 |
| | $3.1\times10^{-2}$ | | Raventos-Duran et al. (2010) | Q | 244 |
| | $6.2\times10^{-2}$ | | Raventos-Duran et al. (2010) | Q | 245 |
| | $5.1\times10^{-2}$ | | Gharagheizi et al. (2010) | Q | 246 |
| | $4.8\times10^{-2}$ | | Hilal et al. (2008) | Q | |
| | $1.3\times10^{-1}$ | | Modarresi et al. (2007) | Q | 67 |
| | | 6900 | Kühne et al. (2005) | Q | |
| | $5.8\times10^{-2}$ | | Yaffe et al. (2003) | Q | 248, 249 |
| | $6.5\times10^{-2}$ | | English and Carroll (2001) | Q | 230, 231 |
| | $1.8\times10^{-2}$ | | Katritzky et al. (1998) | Q | |
| | $7.7\times10^{-2}$ | | Nirmalakhandan et al. (1997) | Q | |
| | | 7800 | Kühne et al. (2005) | ? | |
| | $2.3\times10^{-2}$ | | Yaws et al. (1998) | ? | 38 |
| | $5.6\times10^{-2}$ | | Abraham et al. (1990) | ? | |



Table A3.6: Ketones (RCOR) (...continued)

| Substance Formula (Trivial Name) [CAS Registry Number] InChIKey | $H_s^{cp}$ (at $T^{\ominus}$) $\left[\dfrac{\mathrm{mol}}{\mathrm{m^3\,Pa}}\right]$ | $\dfrac{\mathrm{d}\ln H_s^{cp}}{\mathrm{d}(1/T)}$ [K] | Reference | Type | Note |
|---|---|---|---|---|---|
| 3-methyl-2-hexanone C$_7$H$_{14}$O [2550-21-2] GYWYASONLSQZBB-UHFFFAOYSA-N | $2.8\times10^{-2}$ $4.0\times10^{-3}$ $3.9\times10^{-2}$ $3.2\times10^{-2}$ | | Yaws (2003) Gharagheizi et al. (2012) Gharagheizi et al. (2010) Yaws et al. (1998) | X Q Q ? | 237 246 |
| 4-methyl-2-hexanone C$_7$H$_{14}$O [105-42-0] XUPXMIAWKPTZLZ-UHFFFAOYSA-N | $4.9\times10^{-2}$ $3.6\times10^{-2}$ $9.3\times10^{-2}$ $2.1\times10^{-1}$ $4.0\times10^{-2}$ $3.3\times10^{-2}$ | | Yaws (2003) Wang et al. (2017) Wang et al. (2017) Wang et al. (2017) Gharagheizi et al. (2010) Yaws et al. (1998) | X Q Q Q Q ? | 237 80, 238 80, 239 80, 240 246 |
| 5-methyl-2-hexanone C$_7$H$_{14}$O [110-12-3] FFWSICBKRCICMR-UHFFFAOYSA-N | $6.4\times10^{-2}$ $6.2\times10^{-2}$ $6.2\times10^{-2}$ $4.1\times10^{-2}$ $1.3\times10^{-2}$ $3.6\times10^{-2}$ $1.1\times10^{-1}$ $3.8\times10^{-1}$ $4.0\times10^{-2}$ $7.7\times10^{-2}$ $1.1\times10^{-1}$ $7.3\times10^{-2}$ $2.2\times10^{-2}$ $2.7\times10^{-2}$ | 6800 6900 7600 | Brockbank (2013) Duchowicz et al. (2020) HSDB (2015) Yaws (2003) Duchowicz et al. (2020) Wang et al. (2017) Wang et al. (2017) Wang et al. (2017) Gharagheizi et al. (2010) Hilal et al. (2008) Modarresi et al. (2007) Kühne et al. (2005) Yaffe et al. (2003) Katritzky et al. (1998) Kühne et al. (2005) Yaws et al. (1998) | L V V X Q Q Q Q Q Q Q Q Q Q ? ? | 1 186 237 80, 238 80, 239 80, 240 246 67 248, 249 |
| 2-methyl-3-hexanone C$_7$H$_{14}$O [7379-12-6] HIGGFWFRAWSMBR-UHFFFAOYSA-N | $4.9\times10^{-2}$ $3.7\times10^{-2}$ $4.1\times10^{-2}$ | | Yaws (2003) Gharagheizi et al. (2010) Yaws et al. (1998) | X Q ? | 237 246 |
| 4-methyl-3-hexanone C$_7$H$_{14}$O [17042-16-9] ULPMRIXXHGUZFA-UHFFFAOYSA-N | $3.1\times10^{-2}$ $3.7\times10^{-2}$ $3.7\times10^{-2}$ | | Yaws (2003) Gharagheizi et al. (2010) Yaws et al. (1998) | X Q ? | 237 246 |
| 5-methyl-3-hexanone C$_7$H$_{14}$O [623-56-3] DXVYLFHTJZWTRF-UHFFFAOYSA-N | $3.1\times10^{-2}$ $3.9\times10^{-2}$ $3.7\times10^{-2}$ | | Yaws (2003) Gharagheizi et al. (2010) Yaws et al. (1998) | X Q ? | 237 246 |
| 3-ethyl-2-pentanone C$_7$H$_{14}$O [6137-03-7] GSNKRSKIWFBWEG-UHFFFAOYSA-N | $2.9\times10^{-2}$ $3.9\times10^{-2}$ $3.4\times10^{-2}$ | | Yaws (2003) Gharagheizi et al. (2010) Yaws et al. (1998) | X Q ? | 237 246 |



Table A3.6: Ketones (RCOR) (...continued)

| Substance<br>Formula<br>(Trivial Name)<br>[CAS Registry Number]<br>InChIKey | $H_s^{cp}$<br>(at $T^\ominus$)<br>$\left[\dfrac{\text{mol}}{\text{m}^3\,\text{Pa}}\right]$ | $\dfrac{\text{d}\ln H_s^{cp}}{\text{d}(1/T)}$<br><br>[K] | Reference | Type | Note |
|---|---|---|---|---|---|
| 3,3-dimethyl-2-pentanone<br>$C_7H_{14}O$<br>[20669-04-9]<br>QSHJLBQLQVSEFV-UHFFFAOYSA-N | $3.1\times10^{-2}$<br>$3.3\times10^{-2}$<br>$4.5\times10^{-2}$ | | Yaws (2003)<br>Gharagheizi et al. (2010)<br>Yaws et al. (1998) | X<br>Q<br>? | 237<br>246 |
| 3,4-dimethyl-2-pentanone<br>$C_7H_{14}O$<br>[565-78-6]<br>QXHRQZNDMYRDPA-UHFFFAOYSA-N | $3.2\times10^{-2}$<br>$3.0\times10^{-2}$<br>$4.3\times10^{-2}$ | | Yaws (2003)<br>Gharagheizi et al. (2010)<br>Yaws et al. (1998) | X<br>Q<br>? | 237<br>246 |
| 4,4-dimethyl-2-pentanone<br>$C_7H_{14}O$<br>[590-50-1]<br>AZASWMGVGQEVCS-UHFFFAOYSA-N | $3.6\times10^{-2}$<br>$3.6\times10^{-2}$<br>$5.5\times10^{-2}$ | | Yaws (2003)<br>Gharagheizi et al. (2010)<br>Yaws et al. (1998) | X<br>Q<br>? | 237<br>246 |
| 2,2-dimethyl-3-pentanone<br>$C_7H_{14}O$<br>[564-04-5]<br>VLNUTKMHYLQCQB-UHFFFAOYSA-N | $3.6\times10^{-2}$<br>$3.2\times10^{-2}$<br>$5.5\times10^{-2}$ | | Yaws (2003)<br>Gharagheizi et al. (2010)<br>Yaws et al. (1998) | X<br>Q<br>? | 237<br>246 |
| 2,4-dimethyl-3-pentanone<br>$C_7H_{14}O$<br>(diisopropyl ketone)<br>[565-80-0]<br>HXVNBWAKAOHACI-UHFFFAOYSA-N | $2.3\times10^{-2}$<br>$2.4\times10^{-2}$<br>$2.8\times10^{-2}$<br>$4.1\times10^{-2}$<br>$9.5\times10^{-1}$<br><br>$2.8\times10^{-2}$<br>$4.9\times10^{-3}$<br>$7.8\times10^{-2}$<br>$2.5\times10^{-2}$<br>$6.2\times10^{-2}$<br>$2.5\times10^{-2}$<br>$3.5\times10^{-2}$<br>$1.5\times10^{-1}$<br><br>$2.8\times10^{-2}$<br>$1.4\times10^{-2}$<br>$1.4\times10^{-2}$<br>$1.3\times10^{-2}$<br>$6.0\times10^{-2}$<br><br>$2.8\times10^{-2}$<br>$2.8\times10^{-2}$ | 6100<br>6500<br><br><br>6400<br>6400<br><br><br><br><br><br><br><br><br>6000<br><br><br><br><br><br>4900 | Brockbank (2013)<br>Plyasunov and Shock (2001)<br>Duchowicz et al. (2020)<br>Cabani et al. (1981)<br>Bagno et al. (1991)<br>Della Gatta et al. (1981)<br>Yaws (2003)<br>Duchowicz et al. (2020)<br>Raventos-Duran et al. (2010)<br>Raventos-Duran et al. (2010)<br>Raventos-Duran et al. (2010)<br>Gharagheizi et al. (2010)<br>Hilal et al. (2008)<br>Modarresi et al. (2007)<br>Kühne et al. (2005)<br>Yaffe et al. (2003)<br>Yao et al. (2002)<br>English and Carroll (2001)<br>Katritzky et al. (1998)<br>Nirmalakhandan et al. (1997)<br>Kühne et al. (2005)<br>Yaws (1999)<br>Yaws et al. (1998) | L<br>L<br>V<br>V<br>T<br>T<br>X<br>Q<br>Q<br>Q<br>Q<br>Q<br>Q<br>Q<br>Q<br>Q<br>Q<br>Q<br>Q<br>Q<br>?<br>?<br>? | 1<br><br>186<br><br>473<br><br>237<br><br>271, 243<br>244<br>245<br>246<br><br>67<br><br>248, 249<br>229<br>230, 231<br><br><br><br>21 |
| cycloheptanone<br>$C_7H_{12}O$<br>[502-42-1]<br>CGZZMOTZOONQIA-UHFFFAOYSA-N | $7.0\times10^{-1}$ | | Hilal et al. (2008) | Q | |



Table A3.6: Ketones (RCOR) (...continued)

| Substance Formula (Trivial Name) [CAS Registry Number] InChIKey | $H_s^{cp}$ (at $T^\ominus$) $\left[\dfrac{\text{mol}}{\text{m}^3\,\text{Pa}}\right]$ | $\dfrac{\text{d}\ln H_s^{cp}}{\text{d}(1/T)}$ [K] | Reference | Type | Note |
|---|---|---|---|---|---|
| 2-methylcyclohexanone C$_7$H$_{12}$O [583-60-8] LFSAPCRASZRSKS-UHFFFAOYSA-N | | 5600 4600 | Kühne et al. (2005) Kühne et al. (2005) | Q ? | |
| 3-methylcyclohexanone C$_7$H$_{12}$O [591-24-2] UJBOOUHRTQVGRU-UHFFFAOYSA-N | $6.7\times10^{-2}$ $9.9\times10^{-2}$ | | Duchowicz et al. (2020) Duchowicz et al. (2020) | V Q | 186 |
| 4-methylcyclohexanone C$_7$H$_{12}$O [589-92-4] VGVHNLRUAMRIEW-UHFFFAOYSA-N | | 6500 6100 | Kühne et al. (2005) Kühne et al. (2005) | Q ? | |
| dicyclopropylmethanone C$_7$H$_{10}$O (dicyclopropyl ketone) [1121-37-5] BIPUHAHGLJKIPK-UHFFFAOYSA-N | 3.1 | 7300 7300 | Bagno et al. (1991) Della Gatta et al. (1981) | T T | 473 |
| 2-octanone C$_6$H$_{13}$COCH$_3$ [111-13-7] ZPVFWPFBNIEHGJ-UHFFFAOYSA-N | $5.0\times10^{-2}$ | 7500 | Brockbank (2013) | L | 1, 481 |
| | $5.8\times10^{-2}$ | | Plyasunov and Shock (2001) | L | |
| | $2.9\times10^{-2}$ | | van Ruth et al. (2002) | M | 14 |
| | $3.1\times10^{-2}$ | | van Ruth and Villeneuve (2002) | M | 14, 361 |
| | $1.8\times10^{-2}$ | | van Ruth et al. (2001) | M | 14 |
| | $5.2\times10^{-2}$ | | Buttery et al. (1969) | M | |
| | $4.9\times10^{-2}$ | | Mackay et al. (2006c) | V | |
| | $4.9\times10^{-2}$ | | Mackay et al. (1995) | V | |
| | $5.5\times10^{-2}$ | | Rathbun and Tai (1982) | V | |
| | $7.1\times10^{-2}$ | | Keshavarz et al. (2022) | Q | |
| | $3.2\times10^{-2}$ | | Duchowicz et al. (2020) | Q | 184 |
| | $6.2\times10^{-2}$ | | Raventos-Duran et al. (2010) | Q | 271, 243 |
| | $3.9\times10^{-2}$ | | Raventos-Duran et al. (2010) | Q | 244 |
| | $4.9\times10^{-2}$ | | Raventos-Duran et al. (2010) | Q | 245 |
| | $5.1\times10^{-2}$ | | Hilal et al. (2008) | Q | |
| | $1.2\times10^{-1}$ | | Modarresi et al. (2007) | Q | 67 |
| | | 7300 | Kühne et al. (2005) | Q | |
| | $5.2\times10^{-2}$ | | Yaffe et al. (2003) | Q | 248, 249 |
| | $5.1\times10^{-2}$ | | English and Carroll (2001) | Q | 230, 231 |
| | $2.0\times10^{-2}$ | | Katritzky et al. (1998) | Q | |
| | $5.7\times10^{-2}$ | | Nirmalakhandan et al. (1997) | Q | |
| | $3.3\times10^{-2}$ | | Russell et al. (1992) | Q | 279 |
| | $5.3\times10^{-2}$ | | Suzuki et al. (1992) | Q | 232 |
| | $5.2\times10^{-2}$ | | Duchowicz et al. (2020) | ? | 185, 21 |
| | $5.2\times10^{-2}$ | | Mackay et al. (2006c) | ? | 21 |
| | | 7300 | Kühne et al. (2005) | ? | |
| | $9.6\times10^{-2}$ | | Yaws (1999) | ? | 21, 12 |
| | $1.5\times10^{-1}$ | | Yaws et al. (1998) | ? | 12 |



Table A3.6: Ketones (RCOR) (...continued)

| Substance Formula (Trivial Name) [CAS Registry Number] InChIKey | $H_s^{cp}$ (at $T^{\ominus}$) $\left[\dfrac{\mathrm{mol}}{\mathrm{m^3\,Pa}}\right]$ | $\dfrac{\mathrm{d\ln}H_s^{cp}}{\mathrm{d}(1/T)}$ [K] | Reference | Type | Note |
|---|---|---|---|---|---|
| | $5.2\times10^{-2}$ | | Abraham et al. (1990) | ? | |
| 3-octanone $C_8H_{16}O$ [106-68-3] RHLVCLIPMVJYKS-UHFFFAOYSA-N | $4.3\times10^{-2}$ | 6900 | Brockbank (2013) | L | 1 |
| | $2.9\times10^{-2}$ | 5800 | Wu et al. (2022a) | M | 482 |
| | $7.6\times10^{-2}$ | | HSDB (2015) | V | |
| | $2.5\times10^{-2}$ | | Wang et al. (2017) | Q | 80, 238 |
| | $4.2\times10^{-2}$ | | Wang et al. (2017) | Q | 80, 239 |
| | $1.3\times10^{-1}$ | | Wang et al. (2017) | Q | 80, 240 |
| | $6.2\times10^{-2}$ | | Raventos-Duran et al. (2010) | Q | 271, 243 |
| | $2.5\times10^{-2}$ | | Raventos-Duran et al. (2010) | Q | 244 |
| | $4.9\times10^{-2}$ | | Raventos-Duran et al. (2010) | Q | 245 |
| | $1.1\times10^{-1}$ | | Modarresi et al. (2007) | Q | 67 |
| 4-octanone $C_8H_{16}O$ [589-63-9] YWXLSHOWXZUMSR-UHFFFAOYSA-N | $3.6\times10^{-2}$ | | Hilal et al. (2008) | Q | |
| 5-methyl-3-heptanone $C_8H_{16}O$ [541-85-5] PSBKJPTZCVYXSD-UHFFFAOYSA-N | $5.0\times10^{-2}$ | | Ebert et al. (2023) | ? | 316 |
| 6-methyl-3-heptanone $C_8H_{16}O$ [624-42-0] CCCIYAQYQZQDIZ-UHFFFAOYSA-N | $3.7\times10^{-2}$ | | HSDB (2015) | Q | 99 |
| cyclohexyl methyl ketone $C_6H_{11}COCH_3$ [823-76-7] RIFKADJTWUGDOV-UHFFFAOYSA-N | $2.9\times10^{-1}$ | 7200 | Bagno et al. (1991) | T | 473 |
| | $7.1\times10^{-2}$ | | Keshavarz et al. (2022) | Q | |
| | $9.9\times10^{-2}$ | | Duchowicz et al. (2020) | Q | 184 |
| | $4.1\times10^{-1}$ | | Hilal et al. (2008) | Q | |
| | $4.2\times10^{-1}$ | | Modarresi et al. (2007) | Q | 67 |
| | $2.3\times10^{-1}$ | | English and Carroll (2001) | Q | 230, 231 |
| | $3.1\times10^{-1}$ | | Nirmalakhandan et al. (1997) | Q | |
| | $2.9\times10^{-1}$ | | Duchowicz et al. (2020) | ? | 185, 21 |
| 2-nonanone $C_9H_{18}O$ [821-55-6] VKCYHJWLYTUGCC-UHFFFAOYSA-N | $3.2\times10^{-2}$ | 7400 | Brockbank (2013) | L | 1, 483 |
| | $3.7\times10^{-2}$ | 7500 | Plyasunov and Shock (2001) | L | |
| | $4.1\times10^{-2}$ | | Li and Carr (1993) | M | |
| | $2.7\times10^{-2}$ | | Buttery et al. (1969) | M | |
| | | 7600 | Abraham (1984) | V | |
| | $2.1\times10^{-2}$ | | Yaws (2003) | X | 258 |
| | $1.1\times10^{-1}$ | | Dupeux et al. (2022) | Q | 259 |
| | $9.6\times10^{-2}$ | | Keshavarz et al. (2022) | Q | |
| | $3.2\times10^{-2}$ | | Duchowicz et al. (2020) | Q | 299 |
| | $4.1\times10^{-3}$ | | Gharagheizi et al. (2012) | Q | |
| | $4.9\times10^{-2}$ | | Raventos-Duran et al. (2010) | Q | 242, 243 |
| | $3.9\times10^{-2}$ | | Raventos-Duran et al. (2010) | Q | 244 |
| | $3.9\times10^{-2}$ | | Raventos-Duran et al. (2010) | Q | 245 |





Table A3.6: Ketones (RCOR) (. . . continued)

| Substance<br>Formula<br>(Trivial Name)<br>[CAS Registry Number]<br>InChIKey | $H_s^{cp}$<br>(at $T^{\ominus}$)<br>$\left[\dfrac{\mathrm{mol}}{\mathrm{m^3\,Pa}}\right]$ | $\dfrac{\mathrm{d}\ln H_s^{cp}}{\mathrm{d}(1/T)}$<br><br>[K] | Reference | Type | Note |
|---|---|---|---|---|---|
|  | $4.1\times10^{-2}$ |  | Hilal et al. (2008) | Q |  |
|  | $1.0\times10^{-1}$ |  | Modarresi et al. (2007) | Q | 67 |
|  |  | 7600 | Kühne et al. (2005) | Q |  |
|  | $2.7\times10^{-2}$ |  | Yaffe et al. (2003) | Q | 248, 249 |
|  | $3.9\times10^{-2}$ |  | English and Carroll (2001) | Q | 230, 231 |
|  | $1.3\times10^{-2}$ |  | Katritzky et al. (1998) | Q |  |
|  | $4.4\times10^{-2}$ |  | Nirmalakhandan et al. (1997) | Q |  |
|  | $4.0\times10^{-2}$ |  | Suzuki et al. (1992) | Q | 232 |
|  | $2.7\times10^{-2}$ |  | Duchowicz et al. (2020) | ? | 185, 21 |
|  |  | 8100 | Kühne et al. (2005) | ? |  |
|  | $2.9\times10^{-2}$ |  | Yaws et al. (1998) | ? |  |
|  | $2.7\times10^{-2}$ |  | Abraham et al. (1990) | ? |  |
| 3-nonanone<br>$C_9H_{18}O$<br>[925-78-0]<br>IYTXKIXETAELAV-UHFFFAOYSA-N | $5.1\times10^{-2}$ | 6600 | Brockbank (2013) | L | 1 |
|  | $2.2\times10^{-2}$ |  | Wang et al. (2017) | Q | 80, 238 |
|  | $3.2\times10^{-2}$ |  | Wang et al. (2017) | Q | 80, 239 |
|  | $1.2\times10^{-1}$ |  | Wang et al. (2017) | Q | 80, 240 |
| 5-nonanone<br>$C_9H_{18}O$<br>(dibutyl ketone)<br>[502-56-7]<br>WSGCRAOTEDLMFQ-UHFFFAOYSA-N | $4.0\times10^{-2}$ | 8000 | Brockbank (2013) | L | 1 |
|  | $3.5\times10^{-2}$ |  | Duchowicz et al. (2020) | V | 186 |
|  | $3.5\times10^{-2}$ |  | HSDB (2015) | V |  |
|  | $3.4\times10^{-2}$ |  | Meylan and Howard (1991) | V |  |
|  | $3.7\times10^{-2}$ |  | Cabani et al. (1981) | V |  |
|  | $3.2\times10^{-2}$ |  | Duchowicz et al. (2020) | Q |  |
|  | $2.7\times10^{-2}$ |  | Hilal et al. (2008) | Q |  |
|  | $9.0\times10^{-2}$ |  | Modarresi et al. (2007) | Q | 67 |
|  |  | 7600 | Kühne et al. (2005) | Q |  |
|  | $2.7\times10^{-2}$ |  | Yaffe et al. (2003) | Q | 248, 272 |
|  | $3.9\times10^{-2}$ |  | English and Carroll (2001) | Q | 230, 231 |
|  | $4.7\times10^{-2}$ |  | Nirmalakhandan et al. (1997) | Q |  |
|  | $3.6\times10^{-2}$ |  | Meylan and Howard (1991) | Q |  |
|  |  | 7900 | Kühne et al. (2005) | ? |  |
|  | $3.4\times10^{-2}$ |  | Yaws et al. (1998) | ? | 38 |
|  | $3.5\times10^{-2}$ |  | Abraham et al. (1990) | ? |  |
| 2,6-dimethyl-4-heptanone<br>$C_9H_{18}O$<br>(diisobutyl ketone)<br>[108-83-8]<br>PTTPXKJBFFKCEK-UHFFFAOYSA-N | $1.3\times10^{-2}$ | 7500 | Brockbank (2013) | L | 1 |
|  | $1.3\times10^{-2}$ |  | Plyasunov and Shock (2001) | L |  |
|  | $8.4\times10^{-2}$ |  | Duchowicz et al. (2020) | V | 186 |
|  | $8.2\times10^{-2}$ |  | HSDB (2015) | V |  |
|  | $4.9\times10^{-3}$ |  | Duchowicz et al. (2020) | Q |  |
|  | $3.1\times10^{-2}$ |  | Hilal et al. (2008) | Q |  |
|  | $1.1\times10^{-1}$ |  | Modarresi et al. (2007) | Q | 67 |
|  |  | 7600 | Kühne et al. (2005) | Q |  |
|  | $8.6\times10^{-2}$ |  | Yaffe et al. (2003) | Q | 248, 249 |
|  | $1.2\times10^{-2}$ |  | Katritzky et al. (1998) | Q |  |
|  |  | 5500 | Kühne et al. (2005) | ? |  |
|  | $8.3\times10^{-2}$ |  | Yaws (1999) | ? | 21, 79 |
|  | $9.2\times10^{-2}$ |  | Yaws et al. (1998) | ? | 79 |



Table A3.6: Ketones (RCOR) (...continued)

| Substance Formula (Trivial Name) [CAS Registry Number] InChIKey | $H_s^{cp}$ (at $T^{\ominus}$) $\left[\dfrac{\text{mol}}{\text{m}^3\,\text{Pa}}\right]$ | $\dfrac{\text{d}\ln H_s^{cp}}{\text{d}(1/T)}$ [K] | Reference | Type | Note |
|---|---|---|---|---|---|
| 2,2,4,4-tetramethyl-3-pentanone C$_9$H$_{18}$O (di-(*tert*-butyl) ketone) [815-24-7] UIQGEWJEWJMQSL-UHFFFAOYSA-N | $2.3\times10^{-2}$ | | Bagno et al. (1991) | T | 473 |
| 2-decanone C$_8$H$_{17}$COCH$_3$ [693-54-9] ZAJNGDIORYACQU-UHFFFAOYSA-N | $1.8\times10^{-2}$ | | van Ruth et al. (2002) | M | 14 |
| | $1.5\times10^{-2}$ | | van Ruth and Villeneuve (2002) | M | 14, 361 |
| | $1.5\times10^{-2}$ | | van Ruth et al. (2001) | M | 14 |
| | $2.1\times10^{-2}$ | | Abraham (1984) | V | |
| | $1.7\times10^{-2}$ | | Yaws (2003) | X | 258 |
| | $9.2\times10^{-2}$ | | Dupeux et al. (2022) | Q | 259 |
| | $4.0\times10^{-3}$ | | Gharagheizi et al. (2012) | Q | |
| | $3.4\times10^{-2}$ | | Hilal et al. (2008) | Q | |
| | $8.0\times10^{-2}$ | | Modarresi et al. (2007) | Q | 67 |
| | $2.7\times10^{-2}$ | | Yaffe et al. (2003) | Q | 248, 272 |
| | $3.1\times10^{-2}$ | | English and Carroll (2001) | Q | 230, 260 |
| | $1.4\times10^{-1}$ | | Nirmalakhandan et al. (1997) | Q | |
| | $1.4\times10^{-2}$ | | Yaws et al. (1998) | ? | |
| | $2.1\times10^{-2}$ | | Abraham et al. (1990) | ? | |
| 2-undecanone C$_9$H$_{19}$COCH$_3$ [112-12-9] KYWIYKKSMDLRDC-UHFFFAOYSA-N | $1.9\times10^{-2}$ | | Plyasunov and Shock (2001) | L | |
| | $1.6\times10^{-2}$ | | Buttery et al. (1969) | M | |
| | $1.8\times10^{-2}$ | | Yaws (2003) | X | 258 |
| | $1.7\times10^{-2}$ | | Yaws (2003) | X | 237 |
| | $7.6\times10^{-2}$ | | Dupeux et al. (2022) | Q | 259 |
| | $1.8\times10^{-1}$ | | Keshavarz et al. (2022) | Q | |
| | $3.2\times10^{-2}$ | | Duchowicz et al. (2020) | Q | 299 |
| | $1.5\times10^{-2}$ | | Li et al. (2014) | Q | 241 |
| | $4.1\times10^{-3}$ | | Gharagheizi et al. (2012) | Q | |
| | $2.5\times10^{-2}$ | | Raventos-Duran et al. (2010) | Q | 242, 243 |
| | $4.9\times10^{-1}$ | | Raventos-Duran et al. (2010) | Q | 244 |
| | $2.0\times10^{-2}$ | | Raventos-Duran et al. (2010) | Q | 245 |
| | $2.2\times10^{-2}$ | | Gharagheizi et al. (2010) | Q | 246 |
| | $2.7\times10^{-2}$ | | Hilal et al. (2008) | Q | |
| | $6.3\times10^{-2}$ | | Modarresi et al. (2007) | Q | 67 |
| | $1.6\times10^{-2}$ | | Yaffe et al. (2003) | Q | 248, 249 |
| | $2.3\times10^{-2}$ | | English and Carroll (2001) | Q | 230, 231 |
| | $1.2\times10^{-2}$ | | Katritzky et al. (1998) | Q | |
| | $2.8\times10^{-2}$ | | Nirmalakhandan et al. (1997) | Q | |
| | $2.4\times10^{-2}$ | | Suzuki et al. (1992) | Q | 232 |
| | $1.6\times10^{-1}$ | | Duchowicz et al. (2020) | ? | 185, 21 |
| | $5.8\times10^{-3}$ | | Yaws et al. (1998) | ? | |
| | $1.5\times10^{-2}$ | | Abraham et al. (1990) | ? | |





Table A3.6: Ketones (RCOR) (. . . continued)

| Substance Formula (Trivial Name) [CAS Registry Number] InChIKey | $H_s^{cp}$ (at $T^\ominus$) $\left[\dfrac{\text{mol}}{\text{m}^3\,\text{Pa}}\right]$ | $\dfrac{\mathrm{d}\ln H_s^{cp}}{\mathrm{d}(1/T)}$ [K] | Reference | Type | Note |
|---|---|---|---|---|---|
| 6-undecanone $C_{11}H_{22}O$ [927-49-1] ZPQAKYPOZRXKFA-UHFFFAOYSA-N | $1.5\times10^{-2}$ $6.9\times10^{-2}$ | | Hilal et al. (2008) Modarresi et al. (2007) | Q Q | 67 |
| 2-dodecanone $C_{12}H_{24}O$ [6175-49-1] LSKONYYRONEBKA-UHFFFAOYSA-N | $4.6\times10^{-3}$ $2.1\times10^{-3}$ | | Gharagheizi et al. (2012) Yaws et al. (1998) | Q ? | |
| 2,6,8-trimethyl-4-nonanone $C_{12}H_{24}O$ [123-18-2] GFWVDQCGGDBTBS-UHFFFAOYSA-N | | | Brockbank (2013) | W | 484 |
| 2-tridecanone $C_{13}H_{26}O$ [593-08-8] CYIFVRUOHKNECG-UHFFFAOYSA-N | $4.6\times10^{-3}$ $6.7\times10^{-4}$ | | Gharagheizi et al. (2012) Yaws et al. (1998) | Q ? | |
| 2-tetradecanone $C_{14}H_{28}O$ [2345-27-9] POQLVOYRGNFGRM-UHFFFAOYSA-N | $4.7\times10^{-3}$ $2.1\times10^{-4}$ | | Gharagheizi et al. (2012) Yaws et al. (1998) | Q ? | |
| 2-pentadecanone $C_{15}H_{30}O$ [2345-28-0] CJPNOLIZCWDHJK-UHFFFAOYSA-N | $5.4\times10^{-5}$ | | Yaws et al. (1998) | ? | |
| 2-hexadecanone $C_{16}H_{32}O$ [18787-63-8] XCXKZBWAKKPFCJ-UHFFFAOYSA-N | $1.7\times10^{-5}$ | | Yaws et al. (1998) | ? | |
| 2-heptadecanone $C_{17}H_{34}O$ [2922-51-2] TVTCXPXLRKTHAU-UHFFFAOYSA-N | $3.9\times10^{-6}$ | | Yaws et al. (1998) | ? | |
| menthone $C_{10}H_{18}O$ [89-80-5] NFLGAXVYCFJBMK-UHFFFAOYSA-N | $5.7\times10^{-2}$ $5.0\times10^{-2}$ $6.2\times10^{-2}$ $5.8\times10^{-2}$ | | Marin et al. (1999) Marin et al. (1999) HSDB (2015) Marin et al. (1999) | M V Q Q | 99 |
| tricyclo[3.3.1.1(3,7)]decanone $C_{10}H_{14}O$ (2-adamantanone) [700-58-3] IYKFYARMMIESOX-UHFFFAOYSA-N | 1.4 $7.5\times10^{-1}$ | 5800 | van Roon et al. (2005) Cabani et al. (1981) | V V | |



Table A3.6: Ketones (RCOR) (. . . continued)

| Substance Formula (Trivial Name) [CAS Registry Number] InChIKey | $H_s^{cp}$ (at $T^{\ominus}$) $\left[\dfrac{\text{mol}}{\text{m}^3\,\text{Pa}}\right]$ | $\dfrac{\text{d}\ln H_s^{cp}}{\text{d}(1/T)}$ [K] | Reference | Type | Note |
|---|---|---|---|---|---|
| methyl cedryl ketone $C_{17}H_{26}O$ [32388-55-9] YBUIAJZFOGJGLJ-SWRJLBSHSA-N | 6.1 | | Dupeux et al. (2022) | Q | 259 |
| 3-buten-2-one $C_4H_6O$ (methyl vinyl ketone; MVK) [78-94-4] FUSUHKVFWTUUBE-UHFFFAOYSA-N | $4.0\times10^{-1}$ | | Burkholder et al. (2019) | L | |
| | $4.0\times10^{-1}$ | | Burkholder et al. (2015) | L | |
| | $2.6\times10^{-1}$ | 4800 | Ji and Evans (2007) | M | |
| | $4.0\times10^{-1}$ | | Iraci et al. (1999) | M | |
| | $2.1\times10^{-1}$ | 7800 | Allen et al. (1998) | M | |
| | $1.6\times10^{-1}$ | | Keshavarz et al. (2022) | Q | |
| | $1.1\times10^{-1}$ | | Duchowicz et al. (2020) | Q | 184 |
| | $1.5\times10^{-1}$ | | Wang et al. (2017) | Q | 80, 238 |
| | $3.1\times10^{-1}$ | | Wang et al. (2017) | Q | 80, 239 |
| | $7.8\times10^{-1}$ | | Wang et al. (2017) | Q | 80, 240 |
| | $2.0\times10^{-1}$ | | Raventos-Duran et al. (2010) | Q | 271, 243 |
| | $2.0\times10^{-1}$ | | Raventos-Duran et al. (2010) | Q | 244 |
| | $3.9\times10^{-1}$ | | Raventos-Duran et al. (2010) | Q | 245 |
| | $1.8\times10^{-1}$ | | Hilal et al. (2008) | Q | |
| | $7.1\times10^{-1}$ | | Modarresi et al. (2007) | Q | 67 |
| | | 6000 | Kühne et al. (2005) | Q | |
| | $2.1\times10^{-1}$ | | Duchowicz et al. (2020) | ? | 185, 21 |
| | | 7800 | Kühne et al. (2005) | ? | |
| | $4.3\times10^{-1}$ | | Betterton (1991) | ? | |
| methyl isopropenyl ketone $C_5H_8O$ [814-78-8] ZGHFDIIVVIFNPS-UHFFFAOYSA-N | $9.8\times10^{-2}$ | | Yaws (2003) | X | 237 |
| | $3.0\times10^{-2}$ | | Gharagheizi et al. (2012) | Q | |
| | $9.7\times10^{-2}$ | | Gharagheizi et al. (2010) | Q | 246 |
| | $5.2\times10^{-2}$ | | Yao et al. (2002) | Q | 229 |
| | $9.8\times10^{-2}$ | | Yaws (1999) | ? | 21 |
| 4-methyl-3-penten-2-one $C_6H_{10}O$ (mesityl oxide) [141-79-7] SHOJXDKTYKFBRD-UHFFFAOYSA-N | $2.7\times10^{-1}$ | | HSDB (2015) | V | |
| | $9.9\times10^{-2}$ | | Raventos-Duran et al. (2010) | Q | 271, 243 |
| | $1.6\times10^{-1}$ | | Raventos-Duran et al. (2010) | Q | 244 |
| | $1.6\times10^{-1}$ | | Raventos-Duran et al. (2010) | Q | 245 |
| | $1.8\times10^{-1}$ | | Hilal et al. (2008) | Q | |
| | $2.5\times10^{-1}$ | | Modarresi et al. (2007) | Q | 67 |
| | $2.0\times10^{-1}$ | | Yaws (1999) | ? | 21, 12 |
| 1-phenylethanone $C_6H_5COCH_3$ (acetophenone) [98-86-2] KWOLFJPFCHCOCG-UHFFFAOYSA-N | 1.1 | 7900 | Brockbank (2013) | L | 1 |
| | 1.1 | 7700 | Staudinger and Roberts (2001) | L | |
| | $9.7\times10^{-1}$ | 6800 | Hiatt (2013) | M | |
| | 1.2 | 7800 | Ji et al. (2008) | M | |
| | $9.7\times10^{-1}$ | 12000 | Allen et al. (1998) | M | |
| | $9.3\times10^{-1}$ | | Shiu and Mackay (1997) | M | |
| | 1.1 | 6000 | Betterton (1991) | M | |
| | 1.0 | | Mackay et al. (2006c) | V | |
| | 1.0 | | Shiu and Mackay (1997) | V | |
| | 1.0 | | Mackay et al. (1995) | V | |





Table A3.6: Ketones (RCOR) (…continued)

| Substance Formula (Trivial Name) [CAS Registry Number] InChIKey | $H_s^{cp}$ (at $T^{\ominus}$) $\left[\dfrac{\text{mol}}{\text{m}^3\,\text{Pa}}\right]$ | $\dfrac{\text{d}\ln H_s^{cp}}{\text{d}(1/T)}$ [K] | Reference | Type | Note |
|---|---|---|---|---|---|
| | $9.2\times10^{-1}$ | | Hine and Mookerjee (1975) | V | |
| | $9.5\times10^{-1}$ | 6400 | Bagno et al. (1991) | T | 473 |
| | 1.1 | | Yaws (2003) | X | 258 |
| | $9.3\times10^{-1}$ | | Schüürmann (2000) | C | 21 |
| | 1.8 | | Dupeux et al. (2022) | Q | 259 |
| | $7.1\times10^{-2}$ | | Keshavarz et al. (2022) | Q | |
| | $2.2\times10^{-1}$ | | Duchowicz et al. (2020) | Q | 184 |
| | 2.0 | | Wang et al. (2017) | Q | 80, 238 |
| | 2.0 | | Wang et al. (2017) | Q | 80, 239 |
| | 4.6 | | Wang et al. (2017) | Q | 80, 240 |
| | $9.2\times10^{-1}$ | | Li et al. (2014) | Q | 241 |
| | 1.2 | | Raventos-Duran et al. (2010) | Q | 242, 243 |
| | $9.9\times10^{-1}$ | | Raventos-Duran et al. (2010) | Q | 244 |
| | $9.9\times10^{-1}$ | | Raventos-Duran et al. (2010) | Q | 245 |
| | 1.1 | | Hilal et al. (2008) | Q | |
| | 2.9 | | Modarresi et al. (2007) | Q | 67 |
| | | 6100 | Kühne et al. (2005) | Q | |
| | 1.0 | | Yaffe et al. (2003) | Q | 248, 249 |
| | 1.1 | | English and Carroll (2001) | Q | 230, 231 |
| | $1.9\times10^{-1}$ | | Katritzky et al. (1998) | Q | |
| | $5.3\times10^{-1}$ | | Nirmalakhandan et al. (1997) | Q | |
| | $9.2\times10^{-1}$ | | Suzuki et al. (1992) | Q | 232 |
| | $9.5\times10^{-1}$ | | Duchowicz et al. (2020) | ? | 185, 21 |
| | | 6700 | Kühne et al. (2005) | ? | |
| | 1.1 | | Yaws (1999) | ? | 21 |
| | $9.2\times10^{-1}$ | | Abraham et al. (1990) | ? | |
| 1-phenylethanone-d5 $C_6D_5COCH_3$ (acetophenone-d5) [28077-64-7] KWOLFJPFCHCOCG-VIQYUKPQSA-N | 2.3 | 10000 | Hiatt (2013) | M | |
| 4-methoxy-4-methyl-2-pentanone $C_7H_{14}O_2$ [107-70-0] KOKPBCHLPVDQTK-UHFFFAOYSA-N | 5.1 | | Duchowicz et al. (2020) | V | 186 |
| | 5.1 | | HSDB (2015) | V | |
| | $2.2\times10^{-1}$ | | Duchowicz et al. (2020) | Q | |
| | $7.8\times10^{-1}$ | | Raventos-Duran et al. (2010) | Q | 271, 243 |
| | 2.0 | | Raventos-Duran et al. (2010) | Q | 244 |
| | 7.8 | | Raventos-Duran et al. (2010) | Q | 245 |
| | 1.8 | | Hilal et al. (2008) | Q | |
| | 2.4 | | Modarresi et al. (2007) | Q | 67 |
| 2'-hydroxyacetophenone $C_8H_8O_2$ [118-93-4] JECYUBVRTQDVAT-UHFFFAOYSA-N | 1.0 | 8400 | Ji et al. (2008) | M | |



Table A3.6: Ketones (RCOR) (...continued)

| Substance Formula (Trivial Name) [CAS Registry Number] InChIKey | $H_s^{cp}$ (at $T^\ominus$) $\left[\dfrac{\text{mol}}{\text{m}^3\,\text{Pa}}\right]$ | $\dfrac{\text{d}\ln H_s^{cp}}{\text{d}(1/T)}$ [K] | Reference | Type | Note |
|---|---|---|---|---|---|
| 6-methyl-5-hepten-2-one C$_8$H$_{14}$O [110-93-0] UHEPJGULSIKKTP-UHFFFAOYSA-N | $1.2\times10^{-1}$ $2.5\times10^{-1}$ | 6200 | Wu et al. (2022b) Dupeux et al. (2022) | M Q | 259 |
| phenyl ethyl ketone C$_9$H$_{10}$O (propiophenone) [93-55-0] KRIOVPPHQSLHCZ-UHFFFAOYSA-N | $9.9\times10^{-1}$ $6.6\times10^{-1}$ $7.6\times10^{-2}$ $2.2\times10^{-1}$ 1.9 1.4 $9.1\times10^{-1}$ $7.5\times10^{-1}$ $7.2\times10^{-1}$ 1.6 $9.7\times10^{-1}$ $8.6\times10^{-1}$ $8.0\times10^{-1}$ $7.9\times10^{-2}$ $2.9\times10^{-1}$ | 7900 6400 7700 | Ji et al. (2008) Duchowicz et al. (2020) HSDB (2015) Duchowicz et al. (2020) Wang et al. (2017) Wang et al. (2017) Wang et al. (2017) Zhang et al. (2010) Zhang et al. (2010) Zhang et al. (2010) Zhang et al. (2010) Hilal et al. (2008) Modarresi et al. (2007) Kühne et al. (2005) Yaffe et al. (2003) Katritzky et al. (1998) Kühne et al. (2005) | M V V Q Q Q Q Q Q Q Q Q Q Q Q Q ? | 186 80, 238 80, 239 80, 240 287, 288 287, 289 287, 290 287, 291 67 248, 249 |
| 2′-methylacetophenone C$_9$H$_{10}$O [577-16-2] YXWWHNCQZBVZPV-UHFFFAOYSA-N | $7.6\times10^{-1}$ 1.2 2.4 1.3 | 7600 | Ji et al. (2008) Wang et al. (2017) Wang et al. (2017) Wang et al. (2017) | M Q Q Q | 80, 238 80, 239 80, 240 |
| 3′-methylacetophenone C$_9$H$_{10}$O [585-74-0] FSPSELPMWGWDRY-UHFFFAOYSA-N | 1.5 | 7600 | Ji et al. (2008) | M | |
| 4'-methylacetophenone C$_9$H$_{10}$O [122-00-9] GNKZMNRKLCTJAY-UHFFFAOYSA-N | 1.8 1.1 4.3 1.2 1.2 $9.5\times10^{-1}$ $3.8\times10^{-1}$ | 8400 | Ji et al. (2008) Abraham et al. (1994a) Abney (2021) Hilal et al. (2008) Yaffe et al. (2003) English and Carroll (2001) Nirmalakhandan et al. (1997) | M R Q Q Q Q Q | 399 248, 249 230, 260 |
| 2′-methoxyacetophenone C$_9$H$_{10}$O$_2$ (2-methoxyphenyl methyl ketone) [579-74-8] DWPLEOPKBWNPQV-UHFFFAOYSA-N | $1.7\times10^{1}$ | 9500 | Ji et al. (2008) | M | |





Table A3.6: Ketones (RCOR) (...continued)

| Substance Formula (Trivial Name) [CAS Registry Number] InChIKey | $H_s^{cp}$ (at $T^\ominus$) $\left[\dfrac{\mathrm{mol}}{\mathrm{m}^3\,\mathrm{Pa}}\right]$ | $\dfrac{\mathrm{d}\ln H_s^{cp}}{\mathrm{d}(1/T)}$ [K] | Reference | Type | Note |
|---|---|---|---|---|---|
| 3′-methoxyacetophenone $C_9H_{10}O_2$ (3-methoxyphenyl methyl ketone) [586-37-8] BAYUSCHCCGXLAY-UHFFFAOYSA-N | $1.9\times10^1$ | 9400 | Ji et al. (2008) | M | |
| 4′-methoxyacetophenone $C_9H_{10}O_2$ (4-methoxyphenyl methyl ketone) [100-06-1] NTPLXRHDUXRPNE-UHFFFAOYSA-N | $6.8\times10^{-1}$ 6.9 1.4 1.3 | | Bagno et al. (1991) Hilal et al. (2008) English and Carroll (2001) Nirmalakhandan et al. (1997) | T Q Q Q | 473 230, 231 |
| 2-methyl-5-(1-methylethenyl)-2-cyclohexen-1-one $C_{10}H_{14}O$ (carvone) [6485-40-1] ULDHMXUKGWMISQ-SECBINFHSA-N | $4.9\times10^{-1}$ $5.5\times10^{-1}$ $8.0\times10^{-1}$ | | Amoore and Buttery (1978) Amoore and Buttery (1978) Hilal et al. (2008) | M V Q | |
| benzophenone $C_{13}H_{10}O$ (diphenyl ketone) [119-61-9] RWCCWEUUXYIKHB-UHFFFAOYSA-N | $1.7\times10^1$ 6.1 1.7 9.6 5.2 5.1 2.9 $3.6\times10^1$ $3.4\times10^1$ 1.7 | 9400 | Mackay et al. (2006c) Bagno et al. (1991) Yaws (2003) Dupeux et al. (2022) HSDB (2015) Zhang et al. (2010) Zhang et al. (2010) Zhang et al. (2010) Zhang et al. (2010) Yaws (1999) | V T X Q Q Q Q Q Q ? | 473 258 259 99 287, 288 287, 289 287, 290 287, 291 21 |
| 2,4-dihydroxybenzophenone $C_{13}H_{10}O_3$ [131-56-6] ZXDDPOHVAMWLBH-UHFFFAOYSA-N | $4.6\times10^5$ | | Abraham et al. (2019) | Q | |
| 3,5,5-trimethyl-2-cyclohexen-1-one $C_9H_{14}O$ (isophorone) [78-59-1] HJOVHMDZYOCNQW-UHFFFAOYSA-N | 1.5 1.5 1.7 1.7 1.7 1.7 1.8 $1.7\times10^{-2}$ $2.0\times10^{-1}$ $9.9\times10^{-1}$ $1.6\times10^{-1}$ $6.9\times10^{-1}$ 1.3 | 3900 7300 | Duchowicz et al. (2020) HSDB (2015) Mackay et al. (2006d) Hwang et al. (1992) Suntio et al. (1988) Goldstein (1982) Suntio et al. (1988) Duchowicz et al. (2020) Raventos-Duran et al. (2010) Raventos-Duran et al. (2010) Raventos-Duran et al. (2010) Hilal et al. (2008) Modarresi et al. (2007) Kühne et al. (2005) | V V V V V X C Q Q Q Q Q Q Q | 186 12 298 12 242, 243 244 245 67 |



Table A3.6: Ketones (RCOR) (... continued)

| Substance Formula (Trivial Name) [CAS Registry Number] InChIKey | $H_s^{cp}$ (at $T^\ominus$) $\left[\dfrac{\mathrm{mol}}{\mathrm{m^3\,Pa}}\right]$ | $\dfrac{\mathrm{d\ln} H_s^{cp}}{\mathrm{d}(1/T)}$ [K] | Reference | Type | Note |
|---|---|---|---|---|---|
| | 1.5 | | Yaffe et al. (2003) | Q | 248, 249 |
| | $1.2\times10^{-1}$ | | Katritzky et al. (1998) | Q | |
| | | 7400 | Kühne et al. (2005) | ? | |
| | 1.5 | | Yaws (1999) | ? | 21 |
| bicyclo[2.2.1]heptan-2-one C$_7$H$_{10}$O (norcamphor; 2-norbornanone) [497-38-1] KPMKEVXVVHNIEY-UHFFFAOYSA-N | $4.3\times10^{-1}$ | 5100 | van Roon et al. (2005) | V | |
| 4-methyl-1-(1-methylethyl)- bicyclo[3.1.0]hexan-3-one C$_{10}$H$_{16}$O (thujone) [1125-12-8] USMNOWBWPHYOEA-UHFFFAOYSA-N | $1.0\times10^{-1}$ | 4700 | van Roon et al. (2005) | V | |
| isopropyl phenyl ketone C$_{10}$H$_{12}$O [611-70-1] BSMGLVDZZMBWQB-UHFFFAOYSA-N | $5.7\times10^{-1}$ $3.9\times10^{-1}$ 1.7 $8.2\times10^{-1}$ | | Zhang et al. (2010) Zhang et al. (2010) Zhang et al. (2010) Zhang et al. (2010) | Q Q Q Q | 287, 288 287, 289 287, 290 287, 291 |
| carvone C$_{10}$H$_{14}$O [99-49-0] ULDHMXUKGWMISQ-UHFFFAOYSA-N | $1.3\times10^{-1}$ | | HSDB (2015) | Q | 99 |
| thujone C$_{10}$H$_{16}$O [76231-76-0] USMNOWBWPHYOEA-VWHDNNRLSA-N | $6.2\times10^{-1}$ | | HSDB (2015) | Q | 99 |
| *cis*-jasmone C$_{11}$H$_{16}$O [488-10-8] XMLSXPIVAXONDL-PLNGDYQASA-N | 5.2 | | Dupeux et al. (2022) | Q | 259 |
| $\alpha$-damascone C$_{13}$H$_{20}$O [24720-09-0] CRIGTVCBMUKRSL-FNORWQNLSA-N | $2.1\times10^{1}$ | | Abney (2021) | Q | 399 |
| (E)-$\beta$-damascenone C$_{13}$H$_{18}$O [23726-93-4] POIARNZEYGURDG-FNORWQNLSA-N | $3.2\times10^{-1}$ | 4600 | Wieland et al. (2015) | M | 485 |



Table A3.6: Ketones (RCOR) (...continued)

| Substance Formula (Trivial Name) [CAS Registry Number] InChIKey | $H_s^{cp}$ (at $T^{\ominus}$) $\left[\dfrac{\text{mol}}{\text{m}^3\,\text{Pa}}\right]$ | $\dfrac{\text{d}\ln H_s^{cp}}{\text{d}(1/T)}$ [K] | Reference | Type | Note |
|---|---|---|---|---|---|
| 9H-fluoren-9-one $C_{13}H_8O$ [486-25-9] YLQWCDOCJODRMT-UHFFFAOYSA-N | 6.4 $1.5\times10^1$ $2.3\times10^1$ | | Abraham et al. (2019) HSDB (2015) Parnis et al. (2015) | Q Q Q | 99 369 |
| anthrone $C_{14}H_{10}O$ [90-44-8] RJGDLRCDCYRQOQ-UHFFFAOYSA-N | $1.2\times10^1$ | | HSDB (2015) | Q | 99 |
| 1,2,3,5,6,7-hexahydro-1,1,2,3,3-pentamethyl-4H-inden-4-one $C_{14}H_{22}O$ [33704-61-9] MIZGSAALSYARKU-UHFFFAOYSA-N | $7.0\times10^{-2}$ $6.7\times10^{-3}$ $2.0\times10^1$ $4.8\times10^{-2}$ | | Zhang et al. (2010) Zhang et al. (2010) Zhang et al. (2010) Zhang et al. (2010) | Q Q Q Q | 287, 288 287, 289 287, 290 287, 291 |
| benzoin $C_{14}H_{12}O_2$ [119-53-9] ISAOCJYIOMOJEB-UHFFFAOYSA-N | $2.2\times10^5$ | | Abraham et al. (2019) | Q | |
| methyl-alpha-ionone $C_{14}H_{22}O$ [93302-56-8] VPKMGDRERYMTJX-XEHSLEBBSA-N | 1.2 | | Dupeux et al. (2022) | Q | 259 |
| iso-e super $C_{16}H_{26}O$ [54464-57-2] FVUGZKDGWGKCFE-UHFFFAOYSA-N | $5.5\times10^{-1}$ | | Dupeux et al. (2022) | Q | 259 |
| 2,4,6-trimethylbenzophenone $C_{16}H_{16}O$ [954-16-5] HPAFOABSQZMTHE-UHFFFAOYSA-N | 3.8 3.8 $1.5\times10^1$ 6.0 | | Zhang et al. (2010) Zhang et al. (2010) Zhang et al. (2010) Zhang et al. (2010) | Q Q Q Q | 287, 288 287, 289 287, 290 287, 291 |
| 1-(1,2,3,5,6,7,8,8a-octahydro-2,3,8,8-tetramethyl-2-naphthyl)ethan-1-one $C_{16}H_{26}O$ [68155-66-8] NOMWSTMYQKABST-UHFFFAOYSA-N | $2.5\times10^{-2}$ $3.0\times10^{-1}$ $1.1\times10^1$ $4.0\times10^{-2}$ | | Zhang et al. (2010) Zhang et al. (2010) Zhang et al. (2010) Zhang et al. (2010) | Q Q Q Q | 287, 288 287, 289 287, 290 287, 291 |
| civetone $C_{17}H_{30}O$ [542-46-1] ZKVZSBSZTMPBQR-UPHRSURJSA-N | 3.0 | | Dupeux et al. (2022) | Q | 259 |



Table A3.6: Ketones (RCOR) (... continued)

| Substance<br>Formula<br>(Trivial Name)<br>[CAS Registry Number]<br>InChIKey | $H_s^{cp}$<br>(at $T^\ominus$)<br>$\left[\dfrac{\mathrm{mol}}{\mathrm{m}^3\,\mathrm{Pa}}\right]$ | $\dfrac{\mathrm{d}\ln H_s^{cp}}{\mathrm{d}(1/T)}$<br><br>[K] | Reference | Type | Note |
|---|---|---|---|---|---|
| 1-(2,3-dihydro-1,1,2,3,3,6-<br>hexamethyl-1H-inden-5-<br>yl)ethanone | $3.1{\times}10^{-1}$ | | Zhang et al. (2010) | Q | 287, 288 |
| $C_{17}H_{24}O$ | 2.0 | | Zhang et al. (2010) | Q | 287, 289 |
| [15323-35-0] | 5.2 | | Zhang et al. (2010) | Q | 287, 290 |
| VDBHOHJWUDKDRW-UHFFFAOYSA-N | $9.9{\times}10^{-2}$ | | Zhang et al. (2010) | Q | 287, 291 |
| celestolide | $3.1{\times}10^{-1}$ | | Zhang et al. (2010) | Q | 287, 288 |
| $C_{17}H_{24}O$ | 2.4 | | Zhang et al. (2010) | Q | 287, 289 |
| [13171-00-1] | 3.1 | | Zhang et al. (2010) | Q | 287, 290 |
| IKTHMQYJOWTSJO-UHFFFAOYSA-N | $8.8{\times}10^{-2}$ | | Zhang et al. (2010) | Q | 287, 291 |
| 7H-benz[$de$]anthracen-7-one | $1.5{\times}10^{2}$ | | HSDB (2015) | Q | 99 |
| $C_{17}H_{10}O$ | $8.0{\times}10^{2}$ | | Parnis et al. (2015) | Q | 369 |
| (benzanthrone)<br>[82-05-3]<br>HUKPVYBUJRAUAG-UHFFFAOYSA-N | | | | | |
| 11H-benzo[$a$]fluoren-11-one | $7.2{\times}10^{1}$ | | Parnis et al. (2015) | Q | 369 |
| $C_{17}H_{10}O$<br>[479-79-8]<br>RNICURKFVSAHLQ-UHFFFAOYSA-N | | | | | |
| 11H-benzo[$b$]fluoren-11-one | $5.4{\times}10^{4}$ | | Parnis et al. (2015) | Q | 369 |
| $C_{17}H_{10}O$<br>[3074-03-1]<br>MLMNDNOSVOKYMT-UHFFFAOYSA-N | | | | | |
| benz[$a$]anthracene-7,12-dione | $4.2{\times}10^{2}$ | | Parnis et al. (2015) | Q | 369 |
| $C_{18}H_{10}O_2$<br>[2498-66-0]<br>LHMRXAIRPKSGDE-UHFFFAOYSA-N | | | | | |
| 1-[2,3-dihydro-1,1,2,6-tetramethyl-<br>3-(1-methylethyl)-1H-inden-5-<br>yl]ethanone | $2.3{\times}10^{-1}$ | | Zhang et al. (2010) | Q | 287, 288 |
| $C_{18}H_{26}O$ | 3.2 | | Zhang et al. (2010) | Q | 287, 289 |
| [68140-48-7] | 4.4 | | Zhang et al. (2010) | Q | 287, 290 |
| IMRYETFJNLKUHK-UHFFFAOYSA-N | $1.3{\times}10^{-1}$ | | Zhang et al. (2010) | Q | 287, 291 |
| tonalide | $7.0{\times}10^{-2}$ | | HSDB (2015) | V | |
| $C_{18}H_{26}O$ | $2.3{\times}10^{-1}$ | | Zhang et al. (2010) | Q | 287, 288 |
| [21145-77-7] | 2.4 | | Zhang et al. (2010) | Q | 287, 289 |
| DNRJTBAOUJJKDY-UHFFFAOYSA-N | 7.9 | | Zhang et al. (2010) | Q | 287, 290 |
| | $7.9{\times}10^{-2}$ | | Zhang et al. (2010) | Q | 287, 291 |





Table A3.6: Ketones (RCOR) (...continued)

| Substance Formula (Trivial Name) [CAS Registry Number] InChIKey | $H_s^{cp}$ (at $T^{\ominus}$) $\left[\dfrac{\text{mol}}{\text{m}^3\,\text{Pa}}\right]$ | $\dfrac{\text{d}\ln H_s^{cp}}{\text{d}(1/T)}$ [K] | Reference | Type | Note |
|---|---|---|---|---|---|
| 2,3-butanedione | $7.3\times10^{-1}$ | 5700 | Burkholder et al. (2019) | L | |
| $CH_3COCOCH_3$ | $7.3\times10^{-1}$ | 5700 | Burkholder et al. (2015) | L | |
| (diacetyl; dimethylglyoxal) | $7.3\times10^{-1}$ | 5700 | Sander et al. (2011) | L | |
| [431-03-8] | $1.4\times10^{-1}$ | 4900 | Wu et al. (2022b) | M | |
| QSJXEFYPDANLFS-UHFFFAOYSA-N | $4.2\times10^{-1}$ | 7200 | Wieland et al. (2015) | M | 486 |
| | $5.6\times10^{-1}$ | 6700 | Strekowski and George (2005) | M | |
| | $5.6\times10^{-1}$ | | Straver and de Loos (2005) | M | |
| | $2.1\times10^{-1}$ | | van Ruth et al. (2002) | M | 14 |
| | $2.0\times10^{-1}$ | | van Ruth and Villeneuve (2002) | M | 14, 361 |
| | $2.0\times10^{-1}$ | | van Ruth et al. (2001) | M | 14 |
| | 1.0 | | Marin et al. (1999) | M | |
| | $3.7\times10^{-1}$ | | Roberts and Pollien (1997) | M | |
| | $9.1\times10^{-1}$ | | Landy et al. (1995) | M | |
| | $7.3\times10^{-1}$ | 5700 | Betterton (1991) | M | |
| | $5.7\times10^{-1}$ | | Snider and Dawson (1985) | M | |
| | $6.1\times10^{-1}$ | | Marin et al. (1999) | V | |
| | 1.9 | | Gaffney and Senum (1984) | X | 446 |
| | 1.9 | | Gaffney and Senum (1984) | X | 389 |
| | $1.6\times10^{-1}$ | | Keshavarz et al. (2022) | Q | |
| | 2.2 | | Duchowicz et al. (2020) | Q | 299 |
| | $3.6\times10^{1}$ | | Wang et al. (2017) | Q | 80, 238 |
| | 9.1 | | Wang et al. (2017) | Q | 80, 239 |
| | $1.9\times10^{-1}$ | | Wang et al. (2017) | Q | 80, 240 |
| | $2.0\times10^{1}$ | | Raventos-Duran et al. (2010) | Q | 271, 243 |
| | $1.2\times10^{1}$ | | Raventos-Duran et al. (2010) | Q | 244 |
| | $4.9\times10^{1}$ | | Raventos-Duran et al. (2010) | Q | 245 |
| | 3.8 | | Hilal et al. (2008) | Q | |
| | $1.4\times10^{1}$ | | Modarresi et al. (2007) | Q | 67 |
| | | 6500 | Kühne et al. (2005) | Q | |
| | $7.1\times10^{-1}$ | | Marin et al. (1999) | Q | |
| | $7.4\times10^{-1}$ | | Duchowicz et al. (2020) | ? | 185, 21 |
| | | 6000 | Kühne et al. (2005) | ? | |
| 2,3-pentanedione | $1.3\times10^{-1}$ | 6200 | Wu et al. (2022b) | M | |
| $C_5H_8O_2$ | $3.2\times10^{1}$ | | Wang et al. (2017) | Q | 80, 238 |
| [600-14-6] | 8.7 | | Wang et al. (2017) | Q | 80, 239 |
| TZMFJUDUGYTVRY-UHFFFAOYSA-N | $1.4\times10^{-1}$ | | Wang et al. (2017) | Q | 80, 240 |
| 2,4-pentanedione | $8.5\times10^{-1}$ | 3800 | Brockbank (2013) | L | 1 |
| $C_5H_8O_2$ | 1.5 | | Nozière and Riemer (2003) | M | 79 |
| (acetylacetone) | $9.9\times10^{-1}$ | 4400 | Hovorka et al. (2002) | M | 11 |
| [123-54-6] | 1.7 | | Hellmann (1987) | M | 87 |
| YRKCREAYFQTBPV-UHFFFAOYSA-N | 4.3 | | HSDB (2015) | V | |
| | 3.0 | | Yaws (2003) | X | 237, 12 |
| | $3.2\times10^{1}$ | | Wang et al. (2017) | Q | 80, 238 |
| | $4.5\times10^{1}$ | | Wang et al. (2017) | Q | 80, 239 |
| | $2.5\times10^{1}$ | | Wang et al. (2017) | Q | 80, 240 |
| | $3.9\times10^{1}$ | | Raventos-Duran et al. (2010) | Q | 242, 243 |





Table A3.6: Ketones (RCOR) (...continued)

| Substance Formula (Trivial Name) [CAS Registry Number] InChIKey | $H_s^{cp}$ (at $T^\ominus$) $\left[\dfrac{\mathrm{mol}}{\mathrm{m}^3\,\mathrm{Pa}}\right]$ | $\dfrac{\mathrm{d}\ln H_s^{cp}}{\mathrm{d}(1/T)}$ [K] | Reference | Type | Note |
|---|---|---|---|---|---|
| | $2.5\times10^1$ | | Raventos-Duran et al. (2010) | Q | 244 |
| | $2.0\times10^2$ | | Raventos-Duran et al. (2010) | Q | 245 |
| | 3.2 | | Gharagheizi et al. (2010) | Q | 246 |
| | $1.7\times10^1$ | | Hilal et al. (2008) | Q | |
| | $1.1\times10^1$ | | Modarresi et al. (2007) | Q | 67 |
| | | 7300 | Kühne et al. (2005) | Q | |
| | $1.1\times10^1$ | | Yao et al. (2002) | Q | 229 |
| | | 4400 | Kühne et al. (2005) | ? | |
| | 2.2 | | Yaws (1999) | ? | 21, 12 |
| 1,2-naphthalenedione $C_{10}H_6O_2$ [524-42-5] KETQAJRQOHHATG-UHFFFAOYSA-N | $2.3\times10^3$ | | HSDB (2015) | Q | 99 |
| 1,4-naphthalenedione $C_{10}H_6O_2$ (1,4-naphthoquinone) [130-15-4] FRASJONUBLZVQX-UHFFFAOYSA-N | $5.0\times10^3$ $1.6\times10^2$ | | HSDB (2015) Parnis et al. (2015) | Q Q | 99 369 |
| menadione $C_{11}H_8O_2$ [58-27-5] MJVAVZPDRWSRRC-UHFFFAOYSA-N | $3.2\times10^3$ | | HSDB (2015) | Q | 99 |
| 2,6-di-*tert*-butyl-*p*-benzoquinone $C_{14}H_{20}O_2$ [719-22-2] RDQSIADLBQFVMY-UHFFFAOYSA-N | $6.2\times10^2$ | | HSDB (2015) | Q | 99 |
| 9,10-phenanthrenedione $C_{14}H_8O_2$ [84-11-7] YVYYAPXYZVYDHN-UHFFFAOYSA-N | $3.7\times10^3$ | | HSDB (2015) | Q | 447 |
| 9,10-anthracenedione $C_{14}H_8O_2$ [84-65-1] RZVHIXYEVGDQDX-UHFFFAOYSA-N | $4.2\times10^2$ $4.2\times10^2$ $1.1\times10^{-1}$ $4.9\times10^2$ $1.6\times10^2$ $3.1\times10^3$ $5.6\times10^2$ $1.7\times10^2$ $2.5\times10^4$ $1.1\times10^{-1}$ | | Duchowicz et al. (2020) HSDB (2015) Yaws (2003) Duchowicz et al. (2020) Parnis et al. (2015) Zhang et al. (2010) Zhang et al. (2010) Zhang et al. (2010) Zhang et al. (2010) Gharagheizi et al. (2010) | V V X Q Q Q Q Q Q Q | 186 237 369 287, 288 287, 289 287, 290 287, 291 246 |





Table A3.6: Ketones (RCOR) (...continued)

| Substance / Formula / (Trivial Name) / [CAS Registry Number] / InChIKey | $H_s^{cp}$ (at $T^{\ominus}$) $\left[\dfrac{\mathrm{mol}}{\mathrm{m^3\,Pa}}\right]$ | $\dfrac{\mathrm{d}\ln H_s^{cp}}{\mathrm{d}(1/T)}$ [K] | Reference | Type | Note |
|---|---|---|---|---|---|
| 2-hydroxy-9,10-anthracenedione $C_{14}H_8O_3$ [605-32-3] GCDBEYOJCZLKMC-UHFFFAOYSA-N | $5.4\times10^5$ $2.8\times10^6$ | | Duchowicz et al. (2020) Duchowicz et al. (2020) | V Q | 186 |
| 1,4-dihydroxy-9,10-anthracenedione $C_{14}H_8O_4$ [81-64-1] GUEIZVNYDFNHJU-UHFFFAOYSA-N | $1.2\times10^2$ $1.4\times10^4$ | | Duchowicz et al. (2020) Duchowicz et al. (2020) | V Q | 186 |
| benzil $C_{14}H_{10}O_2$ [134-81-6] WURBFLDFSFBTLW-UHFFFAOYSA-N | $3.0\times10^1$ | | Abraham et al. (2019) | Q | |
| dibenzoylmethane $C_{15}H_{12}O_2$ [120-46-7] NZZIMKJIVMHWJC-UHFFFAOYSA-N | $7.5\times10^3$ $8.0\times10^2$ $6.9\times10^4$ $1.3\times10^4$ | | Zhang et al. (2010) Zhang et al. (2010) Zhang et al. (2010) Zhang et al. (2010) | Q Q Q Q | 287, 288 287, 289 287, 290 287, 291 |
| 2-methyl-9,10-anthracenedione $C_{15}H_{10}O_2$ [84-54-8] NJWGQARXZDRHCD-UHFFFAOYSA-N | $2.4\times10^2$ | | Parnis et al. (2015) | Q | 369 |
| 2-ethyl-9,10-anthracenedione $C_{16}H_{12}O_2$ [84-51-5] SJEBAWHUJDUKQK-UHFFFAOYSA-N | $2.6\times10^1$ $2.1\times10^3$ $4.2\times10^2$ $1.6\times10^2$ $1.1\times10^4$ | | Abraham et al. (2019) Zhang et al. (2010) Zhang et al. (2010) Zhang et al. (2010) Zhang et al. (2010) | Q Q Q Q Q | 287, 288 287, 289 287, 290 287, 291 |
| diphenadione $C_{23}H_{16}O_3$ [82-66-6] JYGLAHSAISAEAL-UHFFFAOYSA-N | $6.4\times10^4$ $2.9\times10^4$ | | Duchowicz et al. (2020) Duchowicz et al. (2020) | V Q | 186 |
| MCM:CH3COCO3H $C_3H_4O_4$ BXASKOSTAOGNPV-UHFFFAOYSA-N | $6.6\times10^4$ $4.7\times10^2$ $4.5\times10^1$ | | Wang et al. (2017) Wang et al. (2017) Wang et al. (2017) | Q Q Q | 80, 238 80, 239 80, 240 |
| MCM:HYPERACET $C_3H_6O_3$ LSIGHSKIPNHVEN-UHFFFAOYSA-N | $5.0\times10^3$ $2.1\times10^2$ $1.4\times10^2$ | | Wang et al. (2017) Wang et al. (2017) Wang et al. (2017) | Q Q Q | 80, 238 80, 239 80, 240 |
| MCM:BIACETOOH $C_4H_6O_4$ JYWVDPVJSGSFMT-UHFFFAOYSA-N | $2.6\times10^6$ $1.2\times10^4$ $5.6\times10^1$ | | Wang et al. (2017) Wang et al. (2017) Wang et al. (2017) | Q Q Q | 80, 238 80, 239 80, 240 |
| MCM:CO2C3CO3H $C_4H_6O_4$ SJOPHMGAZWPERQ-UHFFFAOYSA-N | $5.1\times10^4$ $6.2\times10^3$ $2.8\times10^2$ | | Wang et al. (2017) Wang et al. (2017) Wang et al. (2017) | Q Q Q | 80, 238 80, 239 80, 240 |



Table A3.6: Ketones (RCOR) (...continued)

| Substance Formula (Trivial Name) [CAS Registry Number] InChIKey | $H_s^{cp}$ (at $T^{\ominus}$) $\left[\dfrac{\text{mol}}{\text{m}^3\,\text{Pa}}\right]$ | $\dfrac{\text{d}\ln H_s^{cp}}{\text{d}(1/T)}$ [K] | Reference | Type | Note |
|---|---|---|---|---|---|
| MCM:MEKAOOH | $4.0\times10^3$ | | Wang et al. (2017) | Q | 80, 238 |
| $C_4H_8O_3$ | $6.3\times10^3$ | | Wang et al. (2017) | Q | 80, 239 |
| VEPDQRPXKSSVPR-UHFFFAOYSA-N | $1.6\times10^3$ | | Wang et al. (2017) | Q | 80, 240 |
| MCM:MEKBOOH | $4.7\times10^3$ | | Wang et al. (2017) | Q | 80, 238 |
| $C_4H_8O_3$ | $7.4\times10^1$ | | Wang et al. (2017) | Q | 80, 239 |
| ISENHDUDKGYVAU-UHFFFAOYSA-N | $2.0\times10^2$ | | Wang et al. (2017) | Q | 80, 240 |
| MCM:MEKCOOH | $4.0\times10^3$ | | Wang et al. (2017) | Q | 80, 238 |
| $C_4H_8O_3$ | $1.0\times10^2$ | | Wang et al. (2017) | Q | 80, 239 |
| PSPWSCBXMKHAFW-UHFFFAOYSA-N | $1.4\times10^2$ | | Wang et al. (2017) | Q | 80, 240 |
| MCM:MVKOOH | $1.1\times10^4$ | | Wang et al. (2017) | Q | 80, 238 |
| $C_4H_6O_3$ | $2.6\times10^2$ | | Wang et al. (2017) | Q | 80, 239 |
| PRFKYXWFQJOCGY-UHFFFAOYSA-N | $9.3\times10^1$ | | Wang et al. (2017) | Q | 80, 240 |
| MCM:C41CO3H | $4.8\times10^4$ | | Wang et al. (2017) | Q | 80, 238 |
| $C_5H_8O_4$ | $2.0\times10^3$ | | Wang et al. (2017) | Q | 80, 239 |
| FAYQBJFLFGEPOW-UHFFFAOYSA-N | $3.3$ | | Wang et al. (2017) | Q | 80, 240 |
| MCM:C5CO14OOH | $2.0\times10^5$ | | Wang et al. (2017) | Q | 80, 238 |
| $C_5H_6O_4$ | $7.8\times10^4$ | | Wang et al. (2017) | Q | 80, 239 |
| FKGAAIITVQJZMS-UHFFFAOYSA-N | $4.4$ | | Wang et al. (2017) | Q | 80, 240 |
| MCM:C5CO234 | $2.1\times10^4$ | | Wang et al. (2017) | Q | 80, 238 |
| $C_5H_6O_3$ | $2.2\times10^3$ | | Wang et al. (2017) | Q | 80, 239 |
| MVDYEFQVZNBPPH-UHFFFAOYSA-N | $2.5\times10^{-1}$ | | Wang et al. (2017) | Q | 80, 240 |
| MCM:C5CO234OOH | $1.6\times10^9$ | | Wang et al. (2017) | Q | 80, 238 |
| $C_5H_6O_5$ | $3.0\times10^6$ | | Wang et al. (2017) | Q | 80, 239 |
| XNGUZQHZGCMOHY-UHFFFAOYSA-N | $1.9\times10^1$ | | Wang et al. (2017) | Q | 80, 240 |
| MCM:C5CO23OOH | $2.4\times10^6$ | | Wang et al. (2017) | Q | 80, 238 |
| $C_5H_8O_4$ | $5.3\times10^3$ | | Wang et al. (2017) | Q | 80, 239 |
| BRUGMCFWMGXRAP-UHFFFAOYSA-N | $3.4\times10^1$ | | Wang et al. (2017) | Q | 80, 240 |
| MCM:CO23C4CO3H | $3.2\times10^7$ | | Wang et al. (2017) | Q | 80, 238 |
| $C_5H_6O_5$ | $4.5\times10^5$ | | Wang et al. (2017) | Q | 80, 239 |
| BRGQKBIYWSYIOL-UHFFFAOYSA-N | $4.9\times10^1$ | | Wang et al. (2017) | Q | 80, 240 |
| MCM:CO23C54OOH | $2.5\times10^6$ | | Wang et al. (2017) | Q | 80, 238 |
| $C_5H_8O_4$ | $4.0\times10^3$ | | Wang et al. (2017) | Q | 80, 239 |
| QASRSSCEZIIDDC-UHFFFAOYSA-N | $2.0\times10^1$ | | Wang et al. (2017) | Q | 80, 240 |
| MCM:CO24C53OOH | $2.5\times10^6$ | | Wang et al. (2017) | Q | 80, 238 |
| $C_5H_8O_4$ | $2.1\times10^4$ | | Wang et al. (2017) | Q | 80, 239 |
| AULGAHCEUUOLTF-UHFFFAOYSA-N | $8.7\times10^1$ | | Wang et al. (2017) | Q | 80, 240 |
| MCM:CO2C4CO3H | $1.9\times10^4$ | 12000 | Wieser et al. (2023) | Q | 437 |
| $C_5H_8O_4$ | $4.8\times10^4$ | | Wang et al. (2017) | Q | 80, 238 |
| COVHHGSUFOHLBW-UHFFFAOYSA-N | $1.3\times10^4$ | | Wang et al. (2017) | Q | 80, 239 |
| | $7.1\times10^1$ | | Wang et al. (2017) | Q | 80, 240 |





Table A3.6: Ketones (RCOR) (...continued)

| Substance Formula (Trivial Name) [CAS Registry Number] InChIKey | $H_s^{cp}$ (at $T^\ominus$) $\left[\dfrac{\text{mol}}{\text{m}^3\,\text{Pa}}\right]$ | $\dfrac{\mathrm{d}\ln H_s^{cp}}{\mathrm{d}(1/T)}$ [K] | Reference | Type | Note |
|---|---|---|---|---|---|
| MCM:CO3C4CO3H | $4.8\times10^4$ | | Wang et al. (2017) | Q | 80, 238 |
| $C_5H_8O_4$ | $2.5\times10^3$ | | Wang et al. (2017) | Q | 80, 239 |
| HMOQHHYQXPCOJD-UHFFFAOYSA-N | $1.6\times10^1$ | | Wang et al. (2017) | Q | 80, 240 |
| MCM:DIEKAOOH | $3.7\times10^3$ | | Wang et al. (2017) | Q | 80, 238 |
| $C_5H_{10}O_3$ | $3.6\times10^1$ | | Wang et al. (2017) | Q | 80, 239 |
| GVIFMISHBUBLQF-UHFFFAOYSA-N | $1.0\times10^2$ | | Wang et al. (2017) | Q | 80, 240 |
| MCM:DIEKBOOH | $3.2\times10^3$ | | Wang et al. (2017) | Q | 80, 238 |
| $C_5H_{10}O_3$ | $3.4\times10^3$ | | Wang et al. (2017) | Q | 80, 239 |
| PNIUWVGTVWVIRO-UHFFFAOYSA-N | $3.0\times10^3$ | | Wang et al. (2017) | Q | 80, 240 |
| MCM:MIPKAOOH | $2.6\times10^3$ | | Wang et al. (2017) | Q | 80, 238 |
| $C_5H_{10}O_3$ | $1.6\times10^1$ | | Wang et al. (2017) | Q | 80, 239 |
| UWVYZAHMDOAIIN-UHFFFAOYSA-N | $9.3\times10^1$ | | Wang et al. (2017) | Q | 80, 240 |
| MCM:MIPKBOOH | $3.7\times10^3$ | | Wang et al. (2017) | Q | 80, 238 |
| $C_5H_{10}O_3$ | $3.2\times10^3$ | | Wang et al. (2017) | Q | 80, 239 |
| OWEKNVKRJSTQCB-UHFFFAOYSA-N | $6.8\times10^2$ | | Wang et al. (2017) | Q | 80, 240 |
| MCM:MPRKAOOH | $3.7\times10^3$ | | Wang et al. (2017) | Q | 80, 238 |
| $C_5H_{10}O_3$ | $4.0\times10^1$ | | Wang et al. (2017) | Q | 80, 239 |
| RFXMBEJILJIJHO-UHFFFAOYSA-N | $8.9\times10^1$ | | Wang et al. (2017) | Q | 80, 240 |
| MCM:MPRKBOOH | $3.7\times10^3$ | | Wang et al. (2017) | Q | 80, 238 |
| $C_5H_{10}O_3$ | $2.3\times10^3$ | | Wang et al. (2017) | Q | 80, 239 |
| BOKMIGXVNYNOOE-UHFFFAOYSA-N | $8.5\times10^2$ | | Wang et al. (2017) | Q | 80, 240 |
| MCM:PE2ONE1OOH | $3.2\times10^3$ | | Wang et al. (2017) | Q | 80, 238 |
| $C_5H_{10}O_3$ | $5.8\times10^1$ | | Wang et al. (2017) | Q | 80, 239 |
| NFPIZNOAHNBIPJ-UHFFFAOYSA-N | $3.9\times10^1$ | | Wang et al. (2017) | Q | 80, 240 |
| MCM:C23C54CO3H | $3.0\times10^7$ | | Wang et al. (2017) | Q | 80, 238 |
| $C_6H_8O_5$ | $1.7\times10^5$ | | Wang et al. (2017) | Q | 80, 239 |
| FOQTZZGGVXWRDJ-UHFFFAOYSA-N | $2.0\times10^1$ | | Wang et al. (2017) | Q | 80, 240 |
| MCM:C3COCCO3H | $3.7\times10^4$ | | Wang et al. (2017) | Q | 80, 238 |
| $C_6H_{10}O_4$ | $1.4\times10^3$ | | Wang et al. (2017) | Q | 80, 239 |
| NQEGAXBMSWAEEJ-UHFFFAOYSA-N | $2.6\times10^1$ | | Wang et al. (2017) | Q | 80, 240 |
| MCM:C4DBDIKET | $1.0\times10^2$ | | Wang et al. (2017) | Q | 80, 238 |
| $C_6H_8O_2$ | $1.8\times10^3$ | | Wang et al. (2017) | Q | 80, 239 |
| OTSKZNVDZOOHRX-UHFFFAOYSA-N | $2.6\times10^2$ | | Wang et al. (2017) | Q | 80, 240 |
| MCM:C51CO3H | $4.5\times10^4$ | | Wang et al. (2017) | Q | 80, 238 |
| $C_6H_{10}O_4$ | $3.7\times10^3$ | | Wang et al. (2017) | Q | 80, 239 |
| JNOOBNCRVAKFDO-UHFFFAOYSA-N | $2.5\times10^1$ | | Wang et al. (2017) | Q | 80, 240 |
| MCM:C5CODBCO3H | $1.3\times10^5$ | | Wang et al. (2017) | Q | 80, 238 |
| $C_6H_8O_4$ | $5.6\times10^4$ | | Wang et al. (2017) | Q | 80, 239 |
| DSQMAYIDJPXZOY-UHFFFAOYSA-N | $1.2$ | | Wang et al. (2017) | Q | 80, 240 |





Table A3.6: Ketones (RCOR) (...continued)

| Substance<br>Formula<br>(Trivial Name)<br>[CAS Registry Number]<br>InChIKey | $H_s^{cp}$<br>(at $T^\ominus$)<br>$\left[\dfrac{\text{mol}}{\text{m}^3\,\text{Pa}}\right]$ | $\dfrac{\text{d}\ln H_s^{cp}}{\text{d}(1/T)}$<br><br>[K] | Reference | Type | Note |
|---|---|---|---|---|---|
| MCM:C5DBCOCO3H | $1.3\times10^5$ | | Wang et al. (2017) | Q | 80, 238 |
| $C_6H_8O_4$ | $5.4\times10^4$ | | Wang et al. (2017) | Q | 80, 239 |
| NVTHKHSWEWVMOZ-UHFFFAOYSA-N | 2.2 | | Wang et al. (2017) | Q | 80, 240 |
| MCM:C611OOH | $1.4\times10^6$ | | Wang et al. (2017) | Q | 80, 238 |
| $C_6H_{10}O_4$ | $3.1\times10^3$ | | Wang et al. (2017) | Q | 80, 239 |
| YTLUYDMGHZZGIK-UHFFFAOYSA-N | $1.5\times10^1$ | | Wang et al. (2017) | Q | 80, 240 |
| MCM:C619OOH | $6.0\times10^6$ | | Wang et al. (2017) | Q | 80, 238 |
| $C_6H_8O_4$ | $3.0\times10^6$ | | Wang et al. (2017) | Q | 80, 239 |
| CLPCTPYJHZPKKC-UHFFFAOYSA-N | $1.9\times10^5$ | | Wang et al. (2017) | Q | 80, 240 |
| MCM:C627OOH | $1.9\times10^6$ | | Wang et al. (2017) | Q | 80, 238 |
| $C_6H_{10}O_4$ | $2.3\times10^5$ | | Wang et al. (2017) | Q | 80, 239 |
| BKNNHKNGGIDPKB-UHFFFAOYSA-N | $5.4\times10^3$ | | Wang et al. (2017) | Q | 80, 240 |
| MCM:C62OOH | $1.5\times10^9$ | | Wang et al. (2017) | Q | 80, 238 |
| $C_6H_8O_5$ | $1.4\times10^6$ | | Wang et al. (2017) | Q | 80, 239 |
| NRJLAAIGKSDANL-UHFFFAOYSA-N | $2.5\times10^1$ | | Wang et al. (2017) | Q | 80, 240 |
| MCM:C6CO134OOH | $2.5\times10^7$ | | Wang et al. (2017) | Q | 80, 238 |
| $C_6H_8O_5$ | $2.3\times10^5$ | | Wang et al. (2017) | Q | 80, 239 |
| KXHHBNOARBHHNQ-UHFFFAOYSA-N | $3.7\times10^1$ | | Wang et al. (2017) | Q | 80, 240 |
| MCM:C6CO34 | $2.6\times10^1$ | | Wang et al. (2017) | Q | 80, 238 |
| $C_6H_{10}O_2$ | 5.4 | | Wang et al. (2017) | Q | 80, 239 |
| KVFQMAZOBTXCAZ-UHFFFAOYSA-N | $1.1\times10^{-1}$ | | Wang et al. (2017) | Q | 80, 240 |
| MCM:C6CO34OOH | $2.3\times10^6$ | | Wang et al. (2017) | Q | 80, 238 |
| $C_6H_{10}O_4$ | $2.0\times10^3$ | | Wang et al. (2017) | Q | 80, 239 |
| GOPKJUXJYMJAQC-UHFFFAOYSA-N | $1.5\times10^1$ | | Wang et al. (2017) | Q | 80, 240 |
| MCM:C6CYTONOOH | $4.8\times10^9$ | | Wang et al. (2017) | Q | 80, 238 |
| $C_6H_6O_5$ | $2.1\times10^9$ | | Wang et al. (2017) | Q | 80, 239 |
| JAWAIWJCDBIXQU-UHFFFAOYSA-N | $2.6\times10^7$ | | Wang et al. (2017) | Q | 80, 240 |
| MCM:C6DCRBBOOH | $1.5\times10^5$ | | Wang et al. (2017) | Q | 80, 238 |
| $C_6H_8O_4$ | $2.9\times10^4$ | | Wang et al. (2017) | Q | 80, 239 |
| JRENYXREJDINFO-UHFFFAOYSA-N | 1.4 | | Wang et al. (2017) | Q | 80, 240 |
| MCM:CO234C6 | $1.7\times10^4$ | | Wang et al. (2017) | Q | 80, 238 |
| $C_6H_8O_3$ | $1.1\times10^3$ | | Wang et al. (2017) | Q | 80, 239 |
| FYLKBSXPBXZFJV-UHFFFAOYSA-N | $2.0\times10^{-1}$ | | Wang et al. (2017) | Q | 80, 240 |
| MCM:CO234C6OOH | $1.5\times10^9$ | | Wang et al. (2017) | Q | 80, 238 |
| $C_6H_8O_5$ | $1.1\times10^6$ | | Wang et al. (2017) | Q | 80, 239 |
| RXZSKHWIVPRCNE-UHFFFAOYSA-N | 4.9 | | Wang et al. (2017) | Q | 80, 240 |
| MCM:CO235C6 | $1.7\times10^4$ | | Wang et al. (2017) | Q | 80, 238 |
| $C_6H_8O_3$ | $3.1\times10^3$ | | Wang et al. (2017) | Q | 80, 239 |
| POASQEFMMPWSLD-UHFFFAOYSA-N | $1.6\times10^1$ | | Wang et al. (2017) | Q | 80, 240 |



Table A3.6: Ketones (RCOR) (...continued)

| Substance Formula (Trivial Name) [CAS Registry Number] InChIKey | $H_s^{cp}$ (at $T^{\ominus}$) $\left[\dfrac{\text{mol}}{\text{m}^3\,\text{Pa}}\right]$ | $\dfrac{\text{d}\ln H_s^{cp}}{\text{d}(1/T)}$ [K] | Reference | Type | Note |
|---|---|---|---|---|---|
| MCM:CO235C6OOH | $1.3\times10^9$ | | Wang et al. (2017) | Q | 80, 238 |
| $C_6H_8O_5$ | $2.9\times10^6$ | | Wang et al. (2017) | Q | 80, 239 |
| OFEOQWJKJVUBKA-UHFFFAOYSA-N | $1.2\times10^2$ | | Wang et al. (2017) | Q | 80, 240 |
| MCM:CO23C65OOH | $2.3\times10^6$ | | Wang et al. (2017) | Q | 80, 238 |
| $C_6H_{10}O_4$ | $9.6\times10^4$ | | Wang et al. (2017) | Q | 80, 239 |
| QKJDOFMXAZMTSN-UHFFFAOYSA-N | $3.4\times10^2$ | | Wang et al. (2017) | Q | 80, 240 |
| MCM:CO23C6 | $2.6\times10^1$ | | Wang et al. (2017) | Q | 80, 238 |
| $C_6H_{10}O_2$ | 6.0 | | Wang et al. (2017) | Q | 80, 239 |
| MWVFCEVNXHTDNF-UHFFFAOYSA-N | $7.3\times10^{-2}$ | | Wang et al. (2017) | Q | 80, 240 |
| MCM:CO24C63OOH | $2.3\times10^6$ | | Wang et al. (2017) | Q | 80, 238 |
| $C_6H_{10}O_4$ | $9.1\times10^3$ | | Wang et al. (2017) | Q | 80, 239 |
| BCTNDVKHSPOTQH-UHFFFAOYSA-N | $2.3\times10^1$ | | Wang et al. (2017) | Q | 80, 240 |
| MCM:CO24C6 | $2.6\times10^1$ | | Wang et al. (2017) | Q | 80, 238 |
| $C_6H_{10}O_2$ | $3.5\times10^1$ | | Wang et al. (2017) | Q | 80, 239 |
| NDOGLIPWGGRQCO-UHFFFAOYSA-N | $1.1\times10^1$ | | Wang et al. (2017) | Q | 80, 240 |
| MCM:CO24C6OOH | $2.3\times10^6$ | | Wang et al. (2017) | Q | 80, 238 |
| $C_6H_{10}O_4$ | $1.3\times10^4$ | | Wang et al. (2017) | Q | 80, 239 |
| SLWLMHWJPVVUSV-UHFFFAOYSA-N | $5.8\times10^1$ | | Wang et al. (2017) | Q | 80, 240 |
| MCM:CO24M3C5 | $3.0\times10^1$ | | Wang et al. (2017) | Q | 80, 238 |
| $C_6H_{10}O_2$ | $2.1\times10^1$ | | Wang et al. (2017) | Q | 80, 239 |
| GSOHKPVFCOWKPU-UHFFFAOYSA-N | $1.0\times10^1$ | | Wang et al. (2017) | Q | 80, 240 |
| MCM:CO25C6 | $2.6\times10^1$ | | Wang et al. (2017) | Q | 80, 238 |
| $C_6H_{10}O_2$ | $4.9\times10^2$ | | Wang et al. (2017) | Q | 80, 239 |
| OJVAMHKKJGICOG-UHFFFAOYSA-N | $1.4\times10^2$ | | Wang et al. (2017) | Q | 80, 240 |
| MCM:CO25C6OOH | $2.3\times10^6$ | | Wang et al. (2017) | Q | 80, 238 |
| $C_6H_{10}O_4$ | $6.6\times10^4$ | | Wang et al. (2017) | Q | 80, 239 |
| JXALTZPWVLRWQW-UHFFFAOYSA-N | $6.9\times10^3$ | | Wang et al. (2017) | Q | 80, 240 |
| MCM:CO2C54CO3H | $4.5\times10^4$ | | Wang et al. (2017) | Q | 80, 238 |
| $C_6H_{10}O_4$ | $3.8\times10^3$ | | Wang et al. (2017) | Q | 80, 239 |
| CNZIZHTUCKWXKF-UHFFFAOYSA-N | $4.6\times10^1$ | | Wang et al. (2017) | Q | 80, 240 |
| MCM:CO2M33CO3H | $2.6\times10^4$ | | Wang et al. (2017) | Q | 80, 238 |
| $C_6H_{10}O_4$ | $7.1\times10^2$ | | Wang et al. (2017) | Q | 80, 239 |
| NINAOMADVJMMOS-UHFFFAOYSA-N | 1.3 | | Wang et al. (2017) | Q | 80, 240 |
| MCM:CO3C54CO3H | $4.5\times10^4$ | | Wang et al. (2017) | Q | 80, 238 |
| $C_6H_{10}O_4$ | $8.7\times10^2$ | | Wang et al. (2017) | Q | 80, 239 |
| VXADHVSNPNTBLA-UHFFFAOYSA-N | 1.8 | | Wang et al. (2017) | Q | 80, 240 |
| MCM:CO3C5CO3H | $3.7\times10^4$ | | Wang et al. (2017) | Q | 80, 238 |
| $C_6H_{10}O_4$ | $5.3\times10^3$ | | Wang et al. (2017) | Q | 80, 239 |
| NEAYDZAAQZUWAX-UHFFFAOYSA-N | $3.3\times10^1$ | | Wang et al. (2017) | Q | 80, 240 |



Table A3.6: Ketones (RCOR) (...continued)

| Substance Formula (Trivial Name) [CAS Registry Number] InChIKey | $H_s^{cp}$ (at $T^\ominus$) $\left[\dfrac{\text{mol}}{\text{m}^3\,\text{Pa}}\right]$ | $\dfrac{\text{d}\ln H_s^{cp}}{\text{d}(1/T)}$ [K] | Reference | Type | Note |
|---|---|---|---|---|---|
| MCM:CY6COCOOOH | $6.0\times10^6$ | | Wang et al. (2017) | Q | 80, 238 |
| $C_6H_8O_4$ | $1.1\times10^8$ | | Wang et al. (2017) | Q | 80, 239 |
| PCNCBCRTMCKUHD-UHFFFAOYSA-N | $4.5\times10^7$ | | Wang et al. (2017) | Q | 80, 240 |
| MCM:CY6DIONOOH | $6.0\times10^6$ | | Wang et al. (2017) | Q | 80, 238 |
| $C_6H_8O_4$ | $2.2\times10^7$ | | Wang et al. (2017) | Q | 80, 239 |
| WZMMYHNHRXLTAA-UHFFFAOYSA-N | $5.8\times10^6$ | | Wang et al. (2017) | Q | 80, 240 |
| MCM:CYC613DION | $8.0\times10^1$ | | Wang et al. (2017) | Q | 80, 238 |
| $C_6H_8O_2$ | $2.5\times10^3$ | | Wang et al. (2017) | Q | 80, 239 |
| HJSLFCCWAKVHIW-UHFFFAOYSA-N | $2.6\times10^3$ | | Wang et al. (2017) | Q | 80, 240 |
| MCM:CYC6DIONE | $8.0\times10^1$ | | Wang et al. (2017) | Q | 80, 238 |
| $C_6H_8O_2$ | $6.5\times10^2$ | | Wang et al. (2017) | Q | 80, 239 |
| OILAIQUEIWYQPH-UHFFFAOYSA-N | $8.1\times10^2$ | | Wang et al. (2017) | Q | 80, 240 |
| MCM:CYHXONAOOH | $9.1\times10^3$ | | Wang et al. (2017) | Q | 80, 238 |
| $C_6H_{10}O_3$ | $4.8\times10^4$ | | Wang et al. (2017) | Q | 80, 239 |
| YLRYKXGGQIKRPE-UHFFFAOYSA-N | $5.0\times10^4$ | | Wang et al. (2017) | Q | 80, 240 |
| MCM:ECO3CO3H | $2.0\times10^{10}$ | | Wang et al. (2017) | Q | 80, 238 |
| $C_6H_6O_6$ | $2.6\times10^7$ | | Wang et al. (2017) | Q | 80, 239 |
| LOIDTPBEOYVNRI-UHFFFAOYSA-N | $1.5\times10^2$ | | Wang et al. (2017) | Q | 80, 240 |
| MCM:EIPKAOOH | $2.0\times10^3$ | | Wang et al. (2017) | Q | 80, 238 |
| $C_6H_{12}O_3$ | 8.9 | | Wang et al. (2017) | Q | 80, 239 |
| UUAGRKYHKVSONO-UHFFFAOYSA-N | $4.3\times10^1$ | | Wang et al. (2017) | Q | 80, 240 |
| MCM:EIPKBOOH | $2.9\times10^3$ | | Wang et al. (2017) | Q | 80, 238 |
| $C_6H_{12}O_3$ | $1.7\times10^3$ | | Wang et al. (2017) | Q | 80, 239 |
| GSVDUVPILFIVQC-UHFFFAOYSA-N | $5.4\times10^2$ | | Wang et al. (2017) | Q | 80, 240 |
| MCM:HEX2ONAOOH | $2.9\times10^3$ | | Wang et al. (2017) | Q | 80, 238 |
| $C_6H_{12}O_3$ | $1.4\times10^3$ | | Wang et al. (2017) | Q | 80, 239 |
| BPODHSIMBCVMGL-UHFFFAOYSA-N | $2.9\times10^2$ | | Wang et al. (2017) | Q | 80, 240 |
| MCM:HEX2ONBOOH | $2.9\times10^3$ | | Wang et al. (2017) | Q | 80, 238 |
| $C_6H_{12}O_3$ | $3.5\times10^3$ | | Wang et al. (2017) | Q | 80, 239 |
| XXENGSKOCXRCDY-UHFFFAOYSA-N | $1.3\times10^2$ | | Wang et al. (2017) | Q | 80, 240 |
| MCM:HEX2ONCOOH | $2.9\times10^3$ | | Wang et al. (2017) | Q | 80, 238 |
| $C_6H_{12}O_3$ | $2.8\times10^1$ | | Wang et al. (2017) | Q | 80, 239 |
| UIIPZQPVNWMUDI-UHFFFAOYSA-N | $4.3\times10^1$ | | Wang et al. (2017) | Q | 80, 240 |
| MCM:HEX3ONAOOH | $2.9\times10^3$ | | Wang et al. (2017) | Q | 80, 238 |
| $C_6H_{12}O_3$ | $1.3\times10^3$ | | Wang et al. (2017) | Q | 80, 239 |
| ZKBVMKAKXKAOJC-UHFFFAOYSA-N | $6.2\times10^2$ | | Wang et al. (2017) | Q | 80, 240 |
| MCM:HEX3ONBOOH | $2.9\times10^3$ | | Wang et al. (2017) | Q | 80, 238 |
| $C_6H_{12}O_3$ | $2.2\times10^1$ | | Wang et al. (2017) | Q | 80, 239 |
| CFZVDJFTDOASSJ-UHFFFAOYSA-N | $9.1\times10^1$ | | Wang et al. (2017) | Q | 80, 240 |





Table A3.6: Ketones (RCOR) (...continued)

| Substance<br>Formula<br>(Trivial Name)<br>[CAS Registry Number]<br>InChIKey | $H_s^{cp}$<br>(at $T^\ominus$)<br>$\left[\dfrac{\text{mol}}{\text{m}^3\,\text{Pa}}\right]$ | $\dfrac{\text{d}\ln H_s^{cp}}{\text{d}(1/T)}$<br><br>[K] | Reference | Type | Note |
|---|---|---|---|---|---|
| MCM:HEX3ONCOOH | $2.9\times10^3$ | | Wang et al. (2017) | Q | 80, 238 |
| $C_6H_{12}O_3$ | $2.3\times10^1$ | | Wang et al. (2017) | Q | 80, 239 |
| RCWXBCITQMXQRJ-UHFFFAOYSA-N | $5.4\times10^1$ | | Wang et al. (2017) | Q | 80, 240 |
| MCM:HEX3ONDOOH | $2.5\times10^3$ | | Wang et al. (2017) | Q | 80, 238 |
| $C_6H_{12}O_3$ | $2.2\times10^3$ | | Wang et al. (2017) | Q | 80, 239 |
| GTTLVNDFVPDHBU-UHFFFAOYSA-N | $1.2\times10^3$ | | Wang et al. (2017) | Q | 80, 240 |
| MCM:M2BKAOOH | $3.5\times10^3$ | | Wang et al. (2017) | Q | 80, 238 |
| $C_6H_{12}O_3$ | $1.6\times10^3$ | | Wang et al. (2017) | Q | 80, 239 |
| UYZBTKCJJDJNOE-UHFFFAOYSA-N | $8.0\times10^2$ | | Wang et al. (2017) | Q | 80, 240 |
| MCM:M2BKBOOH | $2.0\times10^3$ | | Wang et al. (2017) | Q | 80, 238 |
| $C_6H_{12}O_3$ | $1.1\times10^1$ | | Wang et al. (2017) | Q | 80, 239 |
| HXTZBJMRLHBTDT-UHFFFAOYSA-N | $3.9\times10^1$ | | Wang et al. (2017) | Q | 80, 240 |
| MCM:MIBK3CO | $3.0\times10^1$ | | Wang et al. (2017) | Q | 80, 238 |
| $C_6H_{10}O_2$ | $5.1$ | | Wang et al. (2017) | Q | 80, 239 |
| JENYBWHRLYZSSZ-UHFFFAOYSA-N | $5.9\times10^{-2}$ | | Wang et al. (2017) | Q | 80, 240 |
| MCM:MIBK3COOOH | $1.4\times10^6$ | | Wang et al. (2017) | Q | 80, 238 |
| $C_6H_{10}O_4$ | $8.9\times10^2$ | | Wang et al. (2017) | Q | 80, 239 |
| REESTLWXXTVHDK-UHFFFAOYSA-N | $6.8$ | | Wang et al. (2017) | Q | 80, 240 |
| MCM:MIBKAOOH | $2.0\times10^3$ | | Wang et al. (2017) | Q | 80, 238 |
| $C_6H_{12}O_3$ | $8.5\times10^2$ | | Wang et al. (2017) | Q | 80, 239 |
| TVLYPTZVJFAYSU-UHFFFAOYSA-N | $1.4\times10^2$ | | Wang et al. (2017) | Q | 80, 240 |
| MCM:MIBKBOOH | $3.5\times10^3$ | | Wang et al. (2017) | Q | 80, 238 |
| $C_6H_{12}O_3$ | $3.2\times10^1$ | | Wang et al. (2017) | Q | 80, 239 |
| NTAUGUQNNYBRPR-UHFFFAOYSA-N | $3.3\times10^1$ | | Wang et al. (2017) | Q | 80, 240 |
| MCM:MTBKOOH | $2.0\times10^3$ | | Wang et al. (2017) | Q | 80, 238 |
| $C_6H_{12}O_3$ | $1.6\times10^3$ | | Wang et al. (2017) | Q | 80, 239 |
| FSDVGLADHCPPED-UHFFFAOYSA-N | $5.1\times10^2$ | | Wang et al. (2017) | Q | 80, 240 |
| MCM:C235C6CO3H | $1.6\times10^{10}$ | | Wang et al. (2017) | Q | 80, 238 |
| $C_7H_8O_6$ | $4.4\times10^7$ | | Wang et al. (2017) | Q | 80, 239 |
| MCULIRAVDBAPRO-UHFFFAOYSA-N | $2.5\times10^2$ | | Wang et al. (2017) | Q | 80, 240 |
| MCM:C710OOH | $1.7\times10^3$ | | Wang et al. (2017) | Q | 80, 238 |
| $C_7H_{14}O_3$ | $5.1\times10^2$ | | Wang et al. (2017) | Q | 80, 239 |
| WMYMHSKADBIKCY-UHFFFAOYSA-N | $6.6\times10^1$ | | Wang et al. (2017) | Q | 80, 240 |
| MCM:C713OOH | $1.2\times10^6$ | | Wang et al. (2017) | Q | 80, 238 |
| $C_7H_{12}O_4$ | $1.2\times10^4$ | | Wang et al. (2017) | Q | 80, 239 |
| BVAIBJLRLSWCNT-UHFFFAOYSA-N | $2.8\times10^2$ | | Wang et al. (2017) | Q | 80, 240 |
| MCM:C714OOH | $2.0\times10^6$ | | Wang et al. (2017) | Q | 80, 238 |
| $C_7H_{12}O_4$ | $3.0\times10^4$ | | Wang et al. (2017) | Q | 80, 239 |
| FQLJODGQMVISEK-UHFFFAOYSA-N | $2.1\times10^3$ | | Wang et al. (2017) | Q | 80, 240 |





Table A3.6: Ketones (RCOR) (...continued)

| Substance Formula (Trivial Name) [CAS Registry Number] InChIKey | $H_s^{cp}$ (at $T^\ominus$) $\left[\dfrac{\text{mol}}{\text{m}^3\,\text{Pa}}\right]$ | $\dfrac{\text{d}\ln H_s^{cp}}{\text{d}(1/T)}$ [K] | Reference | Type | Note |
|---|---|---|---|---|---|
| MCM:C715OOH | $8.3{\times}10^8$ | | Wang et al. (2017) | Q | 80, 238 |
| $C_7H_{10}O_5$ | $2.5{\times}10^5$ | | Wang et al. (2017) | Q | 80, 239 |
| IFAJGMRYFBSYLP-UHFFFAOYSA-N | $2.3{\times}10^1$ | | Wang et al. (2017) | Q | 80, 240 |
| MCM:C7236CO | $6.3{\times}10^4$ | | Wang et al. (2017) | Q | 80, 238 |
| $C_7H_8O_3$ | $6.8{\times}10^4$ | | Wang et al. (2017) | Q | 80, 239 |
| ZXGNNLLPQXKXPQ-UHFFFAOYSA-N | 8.7 | | Wang et al. (2017) | Q | 80, 240 |
| MCM:C726CO5OOH | $7.3{\times}10^6$ | | Wang et al. (2017) | Q | 80, 238 |
| $C_7H_{10}O_4$ | $1.0{\times}10^5$ | | Wang et al. (2017) | Q | 80, 239 |
| DRGOPDAWFSWANX-UHFFFAOYSA-N | $6.8{\times}10^4$ | | Wang et al. (2017) | Q | 80, 240 |
| MCM:C727CO | $1.6{\times}10^4$ | | Wang et al. (2017) | Q | 80, 238 |
| $C_7H_{10}O_3$ | $5.6{\times}10^3$ | | Wang et al. (2017) | Q | 80, 239 |
| XDJAUXXWCBLXGT-UHFFFAOYSA-N | $6.3{\times}10^1$ | | Wang et al. (2017) | Q | 80, 240 |
| MCM:C727OOH | $2.0{\times}10^5$ | 13000 | Wieser et al. (2023) | Q | 437 |
| $C_7H_{12}O_4$ | $1.8{\times}10^6$ | | Wang et al. (2017) | Q | 80, 238 |
| IMCLYNFQESJQRO-UHFFFAOYSA-N | $4.3{\times}10^4$ | | Wang et al. (2017) | Q | 80, 239 |
| | $9.8{\times}10^2$ | | Wang et al. (2017) | Q | 80, 240 |
| MCM:C73OOH | $1.2{\times}10^9$ | | Wang et al. (2017) | Q | 80, 238 |
| $C_7H_{10}O_5$ | $1.1{\times}10^6$ | | Wang et al. (2017) | Q | 80, 239 |
| QFXLTANEYKLVBH-UHFFFAOYSA-N | $7.1{\times}10^1$ | | Wang et al. (2017) | Q | 80, 240 |
| MCM:C74OOH | $1.2{\times}10^9$ | | Wang et al. (2017) | Q | 80, 238 |
| $C_7H_{10}O_5$ | $1.4{\times}10^6$ | | Wang et al. (2017) | Q | 80, 239 |
| ZLQFKUKGFNRBMM-UHFFFAOYSA-N | $4.8{\times}10^2$ | | Wang et al. (2017) | Q | 80, 240 |
| MCM:C75OOH | $2.7{\times}10^3$ | | Wang et al. (2017) | Q | 80, 238 |
| $C_7H_{14}O_3$ | $8.9{\times}10^2$ | | Wang et al. (2017) | Q | 80, 239 |
| YNKFHYFADYHYPZ-UHFFFAOYSA-N | $1.8{\times}10^2$ | | Wang et al. (2017) | Q | 80, 240 |
| MCM:C7ADCCO3H | $8.5{\times}10^4$ | | Wang et al. (2017) | Q | 80, 238 |
| $C_7H_{10}O_4$ | $6.5{\times}10^4$ | | Wang et al. (2017) | Q | 80, 239 |
| KFHCVBUEDOHUEG-UHFFFAOYSA-N | 5.0 | | Wang et al. (2017) | Q | 80, 240 |
| MCM:C7BDICARB | $6.9{\times}10^1$ | | Wang et al. (2017) | Q | 80, 238 |
| $C_7H_{10}O_2$ | $1.4{\times}10^3$ | | Wang et al. (2017) | Q | 80, 239 |
| SGWFXGZCQBUONH-UHFFFAOYSA-N | $3.9{\times}10^1$ | | Wang et al. (2017) | Q | 80, 240 |
| MCM:C7DCCO3H | $1.4{\times}10^5$ | | Wang et al. (2017) | Q | 80, 238 |
| $C_7H_{10}O_4$ | $1.5{\times}10^4$ | | Wang et al. (2017) | Q | 80, 239 |
| VBFDFNFUMYAFQG-UHFFFAOYSA-N | 1.5 | | Wang et al. (2017) | Q | 80, 240 |
| MCM:C7DDCCO3H | $1.0{\times}10^5$ | | Wang et al. (2017) | Q | 80, 238 |
| $C_7H_{10}O_4$ | $2.2{\times}10^4$ | | Wang et al. (2017) | Q | 80, 239 |
| ITRURPAVBUVUCD-UHFFFAOYSA-N | $5.6{\times}10^{-1}$ | | Wang et al. (2017) | Q | 80, 240 |
| MCM:C7EDICARB | $9.3{\times}10^1$ | | Wang et al. (2017) | Q | 80, 238 |
| $C_7H_{10}O_2$ | $1.1{\times}10^3$ | | Wang et al. (2017) | Q | 80, 239 |
| YTVSAHBJOWKZHT-UHFFFAOYSA-N | $1.6{\times}10^2$ | | Wang et al. (2017) | Q | 80, 240 |



Table A3.6: Ketones (RCOR) (...continued)

| Substance Formula (Trivial Name) [CAS Registry Number] InChIKey | $H_s^{cp}$ (at $T^{\ominus}$) $\left[\dfrac{\text{mol}}{\text{m}^3\,\text{Pa}}\right]$ | $\dfrac{\text{d}\ln H_s^{cp}}{\text{d}(1/T)}$ [K] | Reference | Type | Note |
|---|---|---|---|---|---|
| MCM:C821OOH | $1.2\times10^9$ | | Wang et al. (2017) | Q | 80, 238 |
| $C_7H_{10}O_5$ | $2.6\times10^6$ | | Wang et al. (2017) | Q | 80, 239 |
| VNKKTGYQOPWEKF-UHFFFAOYSA-N | $3.2\times10^2$ | | Wang et al. (2017) | Q | 80, 240 |
| MCM:CO235C7 | $1.6\times10^4$ | | Wang et al. (2017) | Q | 80, 238 |
| $C_7H_{10}O_3$ | $9.6\times10^2$ | | Wang et al. (2017) | Q | 80, 239 |
| PLNIROICNOROEN-UHFFFAOYSA-N | 9.6 | | Wang et al. (2017) | Q | 80, 240 |
| MCM:CO245C7 | $1.6\times10^4$ | | Wang et al. (2017) | Q | 80, 238 |
| $C_7H_{10}O_3$ | $9.6\times10^2$ | | Wang et al. (2017) | Q | 80, 239 |
| SOSHREJUCDQQIR-UHFFFAOYSA-N | $1.1\times10^1$ | | Wang et al. (2017) | Q | 80, 240 |
| MCM:CO25C6CO3H | $2.3\times10^7$ | | Wang et al. (2017) | Q | 80, 238 |
| $C_7H_{10}O_5$ | $4.6\times10^6$ | | Wang et al. (2017) | Q | 80, 239 |
| NZAOVWOYFADSGM-UHFFFAOYSA-N | $7.4\times10^3$ | | Wang et al. (2017) | Q | 80, 240 |
| MCM:CO25C73OOH | $1.8\times10^6$ | | Wang et al. (2017) | Q | 80, 238 |
| $C_7H_{12}O_4$ | $2.8\times10^4$ | | Wang et al. (2017) | Q | 80, 239 |
| NWTMJMDPQCUZCE-UHFFFAOYSA-N | $9.1\times10^2$ | | Wang et al. (2017) | Q | 80, 240 |
| MCM:CO25C74OOH | $1.8\times10^6$ | | Wang et al. (2017) | Q | 80, 238 |
| $C_7H_{12}O_4$ | $2.8\times10^4$ | | Wang et al. (2017) | Q | 80, 239 |
| VDRXHYKLZSMLBZ-UHFFFAOYSA-N | $3.6\times10^3$ | | Wang et al. (2017) | Q | 80, 240 |
| MCM:CO25C7 | $2.0\times10^1$ | | Wang et al. (2017) | Q | 80, 238 |
| $C_7H_{12}O_2$ | $2.6\times10^2$ | | Wang et al. (2017) | Q | 80, 239 |
| HGRGPAAXHOTBAM-UHFFFAOYSA-N | $8.3\times10^1$ | | Wang et al. (2017) | Q | 80, 240 |
| MCM:CO3C75OOH | $2.7\times10^3$ | | Wang et al. (2017) | Q | 80, 238 |
| $C_7H_{14}O_3$ | $8.0\times10^2$ | | Wang et al. (2017) | Q | 80, 239 |
| TVVPPVSOZMPRIM-UHFFFAOYSA-N | $4.3\times10^2$ | | Wang et al. (2017) | Q | 80, 240 |
| MCM:IC7DCCO3H | $1.4\times10^5$ | | Wang et al. (2017) | Q | 80, 238 |
| $C_7H_{10}O_4$ | $1.6\times10^4$ | | Wang et al. (2017) | Q | 80, 239 |
| JFPQYVJBAXIOAT-UHFFFAOYSA-N | $1.9\times10^1$ | | Wang et al. (2017) | Q | 80, 240 |
| MCM:M3CO245C6 | $1.6\times10^4$ | | Wang et al. (2017) | Q | 80, 238 |
| $C_7H_{10}O_3$ | $6.3\times10^2$ | | Wang et al. (2017) | Q | 80, 239 |
| FMABCBYNHKBBEH-UHFFFAOYSA-N | 8.5 | | Wang et al. (2017) | Q | 80, 240 |
| MCM:M3CO25C6 | $2.3\times10^1$ | | Wang et al. (2017) | Q | 80, 238 |
| $C_7H_{12}O_2$ | $1.8\times10^2$ | | Wang et al. (2017) | Q | 80, 239 |
| NLLILAUVOOREKR-UHFFFAOYSA-N | $8.5\times10^1$ | | Wang et al. (2017) | Q | 80, 240 |
| MCM:C5DBECO3H | $6.0\times10^7$ | | Wang et al. (2017) | Q | 80, 238 |
| $C_8H_{10}O_5$ | $1.5\times10^6$ | | Wang et al. (2017) | Q | 80, 239 |
| QHSWUEYXPZRIAH-UHFFFAOYSA-N | 1.1 | | Wang et al. (2017) | Q | 80, 240 |
| MCM:C5EDBCO3H | $6.0\times10^7$ | | Wang et al. (2017) | Q | 80, 238 |
| $C_8H_{10}O_5$ | $1.6\times10^6$ | | Wang et al. (2017) | Q | 80, 239 |
| YXJCVNQUVAHMKT-UHFFFAOYSA-N | $6.9\times10^{-1}$ | | Wang et al. (2017) | Q | 80, 240 |



Table A3.6: Ketones (RCOR) (...continued)

| Substance / Formula / (Trivial Name) / [CAS Registry Number] / InChIKey | $H_s^{cp}$ (at $T^{\ominus}$) $\left[\dfrac{\text{mol}}{\text{m}^3\,\text{Pa}}\right]$ | $\dfrac{\mathrm{d}\ln H_s^{cp}}{\mathrm{d}(1/T)}$ [K] | Reference | Type | Note |
|---|---|---|---|---|---|
| MCM:C727CO3H | $2.1\times10^7$ | | Wang et al. (2017) | Q | 80, 238 |
| $C_8H_{12}O_5$ | $9.6\times10^5$ | | Wang et al. (2017) | Q | 80, 239 |
| FJTLPXSURBYIDW-UHFFFAOYSA-N | $1.3\times10^3$ | | Wang et al. (2017) | Q | 80, 240 |
| MCM:C7CO2M5OOH | $4.0\times10^6$ | | Wang et al. (2017) | Q | 80, 238 |
| $C_8H_{12}O_4$ | $1.1\times10^4$ | | Wang et al. (2017) | Q | 80, 239 |
| GOMWBXPFFZKXIF-UHFFFAOYSA-N | $2.3\times10^4$ | | Wang et al. (2017) | Q | 80, 240 |
| MCM:C7M2CO5OOH | $4.9\times10^6$ | | Wang et al. (2017) | Q | 80, 238 |
| $C_8H_{12}O_4$ | $7.6\times10^4$ | | Wang et al. (2017) | Q | 80, 239 |
| BVDWDBGMQXZSJX-UHFFFAOYSA-N | $4.0\times10^4$ | | Wang et al. (2017) | Q | 80, 240 |
| MCM:C7M3CO | $4.3\times10^4$ | | Wang et al. (2017) | Q | 80, 238 |
| $C_8H_{10}O_3$ | $8.1\times10^4$ | | Wang et al. (2017) | Q | 80, 239 |
| WFGYJTNUNWJHJN-UHFFFAOYSA-N | $2.3\times10^1$ | | Wang et al. (2017) | Q | 80, 240 |
| MCM:C7ODLBCO3H | $8.1\times10^4$ | | Wang et al. (2017) | Q | 80, 238 |
| $C_8H_{12}O_4$ | $1.2\times10^4$ | | Wang et al. (2017) | Q | 80, 239 |
| NMESNSUDSMNPLZ-UHFFFAOYSA-N | $3.2\times10^{-1}$ | | Wang et al. (2017) | Q | 80, 240 |
| MCM:C816CO | $3.1\times10^2$ | 10000 | Wieser et al. (2023) | Q | 437 |
| $C_8H_{12}O_2$ | $3.8\times10^1$ | | Wang et al. (2017) | Q | 80, 238 |
| FXXXYZSMIIVJDG-UHFFFAOYSA-N | $3.0\times10^2$ | | Wang et al. (2017) | Q | 80, 239 |
| | $6.2\times10^1$ | | Wang et al. (2017) | Q | 80, 240 |
| MCM:C816OOH | $4.9\times10^3$ | | Wang et al. (2017) | Q | 80, 238 |
| $C_8H_{14}O_3$ | $2.0\times10^3$ | | Wang et al. (2017) | Q | 80, 239 |
| GWYFGLCLLKWWIQ-UHFFFAOYSA-N | $1.7\times10^2$ | | Wang et al. (2017) | Q | 80, 240 |
| MCM:C817OOH | $1.2\times10^6$ | 13000 | Wieser et al. (2023) | Q | 437 |
| $C_8H_{14}O_4$ | $1.5\times10^6$ | | Wang et al. (2017) | Q | 80, 238 |
| WQMXPSRLZZCZTO-UHFFFAOYSA-N | $1.5\times10^6$ | | Wang et al. (2017) | Q | 80, 239 |
| | $1.6\times10^4$ | | Wang et al. (2017) | Q | 80, 240 |
| MCM:C8236CO | $4.9\times10^4$ | | Wang et al. (2017) | Q | 80, 238 |
| $C_8H_{10}O_3$ | $6.0\times10^4$ | | Wang et al. (2017) | Q | 80, 239 |
| IOHUPZZTKNTURH-UHFFFAOYSA-N | $4.8$ | | Wang et al. (2017) | Q | 80, 240 |
| MCM:C826CO3OOH | $6.5\times10^6$ | | Wang et al. (2017) | Q | 80, 238 |
| $C_8H_{12}O_4$ | $4.4\times10^4$ | | Wang et al. (2017) | Q | 80, 239 |
| JGDMXPACXAYITG-UHFFFAOYSA-N | $1.6\times10^4$ | | Wang et al. (2017) | Q | 80, 240 |
| MCM:C827OOH | $9.8\times10^5$ | | Wang et al. (2017) | Q | 80, 238 |
| $C_8H_{14}O_4$ | $1.5\times10^4$ | | Wang et al. (2017) | Q | 80, 239 |
| MYZHYUNBFYQPIX-UHFFFAOYSA-N | $1.5\times10^3$ | | Wang et al. (2017) | Q | 80, 240 |
| MCM:C828OOH | $6.8\times10^8$ | | Wang et al. (2017) | Q | 80, 238 |
| $C_8H_{12}O_5$ | $3.1\times10^5$ | | Wang et al. (2017) | Q | 80, 239 |
| IPIPKWBUGMDLTO-UHFFFAOYSA-N | $2.5\times10^1$ | | Wang et al. (2017) | Q | 80, 240 |
| MCM:C83OOH | $1.1\times10^9$ | | Wang et al. (2017) | Q | 80, 238 |
| $C_8H_{12}O_5$ | $3.6\times10^5$ | | Wang et al. (2017) | Q | 80, 239 |
| MTDHYROWVQXAOV-UHFFFAOYSA-N | $5.6$ | | Wang et al. (2017) | Q | 80, 240 |



Table A3.6: Ketones (RCOR) (...continued)

| Substance Formula (Trivial Name) [CAS Registry Number] InChIKey | $H_s^{cp}$ (at $T^{\ominus}$) $\left[\dfrac{\mathrm{mol}}{\mathrm{m^3\,Pa}}\right]$ | $\dfrac{\mathrm{d}\ln H_s^{cp}}{\mathrm{d}(1/T)}$ [K] | Reference | Type | Note |
|---|---|---|---|---|---|
| MCM:C84OOH $C_8H_{14}O_4$ HQUQRVBRCVTTSE-UHFFFAOYSA-N | $1.5\times10^6$ $1.4\times10^4$ $7.1\times10^2$ | | Wang et al. (2017) Wang et al. (2017) Wang et al. (2017) | Q Q Q | 80, 238 80, 239 80, 240 |
| MCM:C85OOH $C_8H_{14}O_3$ MGHQHHXZPXRHRQ-UHFFFAOYSA-N | $5.5\times10^3$ $5.5\times10^3$ $1.0\times10^4$ | | Wang et al. (2017) Wang et al. (2017) Wang et al. (2017) | Q Q Q | 80, 238 80, 239 80, 240 |
| MCM:C88CO $C_8H_{10}O_3$ KYJUTSWXKQCOOZ-UHFFFAOYSA-N | $2.8\times10^4$ $7.4\times10^4$ $3.3\times10^3$ | | Wang et al. (2017) Wang et al. (2017) Wang et al. (2017) | Q Q Q | 80, 238 80, 239 80, 240 |
| MCM:C88OOH $C_8H_{12}O_4$ LSPOECQNSUKNPK-UHFFFAOYSA-N | $3.2\times10^6$ $6.9\times10^5$ $1.4\times10^4$ | | Wang et al. (2017) Wang et al. (2017) Wang et al. (2017) | Q Q Q | 80, 238 80, 239 80, 240 |
| MCM:C8BCCO $C_8H_{12}O$ BYOKRVKHSHTAOM-UHFFFAOYSA-N | $2.0\times10^{-1}$ $9.6\times10^{-1}$ $3.7$ | | Wang et al. (2017) Wang et al. (2017) Wang et al. (2017) | Q Q Q | 80, 238 80, 239 80, 240 |
| MCM:CO346C8 $C_8H_{12}O_3$ YUHNMHTZZXLCQZ-UHFFFAOYSA-N | $1.3\times10^4$ $1.1\times10^3$ $7.3$ | | Wang et al. (2017) Wang et al. (2017) Wang et al. (2017) | Q Q Q | 80, 238 80, 239 80, 240 |
| MCM:CO36C8 $C_8H_{14}O_2$ CVZGUJMLZZTPKH-UHFFFAOYSA-N | $1.8\times10^1$ $1.5\times10^2$ $3.6\times10^1$ | | Wang et al. (2017) Wang et al. (2017) Wang et al. (2017) | Q Q Q | 80, 238 80, 239 80, 240 |
| MCM:CO3C85OOH $C_8H_{16}O_3$ RPACFUNWVVDXHA-UHFFFAOYSA-N | $2.1\times10^3$ $5.9\times10^2$ $3.0\times10^2$ | | Wang et al. (2017) Wang et al. (2017) Wang et al. (2017) | Q Q Q | 80, 238 80, 239 80, 240 |
| MCM:MXYQONE $C_8H_8O_2$ SENUUPBBLQWHMF-UHFFFAOYSA-N | $2.1$ $5.3\times10^3$ $1.8\times10^1$ | | Wang et al. (2017) Wang et al. (2017) Wang et al. (2017) | Q Q Q | 80, 238 80, 239 80, 240 |
| MCM:OXYQONE $C_8H_8O_2$ AIACLXROWHONEE-UHFFFAOYSA-N | $2.1$ $5.3\times10^3$ $1.0\times10^1$ | | Wang et al. (2017) Wang et al. (2017) Wang et al. (2017) | Q Q Q | 80, 238 80, 239 80, 240 |
| MCM:PEBQONE $C_8H_8O_2$ IGRSQEOIAAGSGS-UHFFFAOYSA-N | $2.7$ $4.7\times10^3$ $1.7\times10^1$ | | Wang et al. (2017) Wang et al. (2017) Wang et al. (2017) | Q Q Q | 80, 238 80, 239 80, 240 |
| MCM:PXYQONE $C_8H_8O_2$ MYKLQMNSFPAPLZ-UHFFFAOYSA-N | $2.1$ $5.3\times10^3$ $1.6\times10^1$ | | Wang et al. (2017) Wang et al. (2017) Wang et al. (2017) | Q Q Q | 80, 238 80, 239 80, 240 |
| MCM:C816CO3H $C_9H_{14}O_4$ IXOBJDNIWKDFSA-UHFFFAOYSA-N | $5.3\times10^4$ $1.5\times10^3$ $3.9\times10^1$ | | Wang et al. (2017) Wang et al. (2017) Wang et al. (2017) | Q Q Q | 80, 238 80, 239 80, 240 |



Table A3.6: Ketones (RCOR) (...continued)

| Substance Formula (Trivial Name) [CAS Registry Number] InChIKey | $H_s^{cp}$ (at $T^\ominus$) $\left[\dfrac{\mathrm{mol}}{\mathrm{m^3\,Pa}}\right]$ | $\dfrac{\mathrm{d}\ln H_s^{cp}}{\mathrm{d}(1/T)}$ [K] | Reference | Type | Note |
|---|---|---|---|---|---|
| MCM:C817CO3H | $3.4\times10^7$ | 16000 | Wieser et al. (2023) | Q | 437 |
| $C_9H_{14}O_5$ | $1.7\times10^7$ | | Wang et al. (2017) | Q | 80, 238 |
| NVOUBBWYYUIRJW-UHFFFAOYSA-N | $1.7\times10^6$ | | Wang et al. (2017) | Q | 80, 239 |
| | $2.0\times10^4$ | | Wang et al. (2017) | Q | 80, 240 |
| MCM:C827CO3H | $1.2\times10^7$ | | Wang et al. (2017) | Q | 80, 238 |
| $C_9H_{14}O_5$ | $6.0\times10^5$ | | Wang et al. (2017) | Q | 80, 239 |
| DZJUGPICXPPSEN-UHFFFAOYSA-N | $1.4\times10^2$ | | Wang et al. (2017) | Q | 80, 240 |
| MCM:C828CO3H | $7.8\times10^9$ | | Wang et al. (2017) | Q | 80, 238 |
| $C_9H_{12}O_6$ | $2.0\times10^7$ | | Wang et al. (2017) | Q | 80, 239 |
| XCCOWFUPSKLDCA-UHFFFAOYSA-N | $8.9\times10^1$ | | Wang et al. (2017) | Q | 80, 240 |
| MCM:C85CO3H | $5.8\times10^4$ | | Wang et al. (2017) | Q | 80, 238 |
| $C_9H_{14}O_4$ | $3.8\times10^3$ | | Wang et al. (2017) | Q | 80, 239 |
| DAYKQISVSRMABH-UHFFFAOYSA-N | $1.1\times10^2$ | | Wang et al. (2017) | Q | 80, 240 |
| MCM:C88CO3H | $3.7\times10^7$ | | Wang et al. (2017) | Q | 80, 238 |
| $C_9H_{12}O_5$ | $2.0\times10^7$ | | Wang et al. (2017) | Q | 80, 239 |
| KOEDWBONHQRVNE-UHFFFAOYSA-N | $4.8\times10^4$ | | Wang et al. (2017) | Q | 80, 240 |
| MCM:C8M2CO6OOH | $3.9\times10^6$ | | Wang et al. (2017) | Q | 80, 238 |
| $C_9H_{14}O_4$ | $3.3\times10^4$ | | Wang et al. (2017) | Q | 80, 239 |
| LMHVWEUXUKAUEE-UHFFFAOYSA-N | $1.0\times10^4$ | | Wang et al. (2017) | Q | 80, 240 |
| MCM:C8M3CO | $3.3\times10^4$ | | Wang et al. (2017) | Q | 80, 238 |
| $C_9H_{12}O_3$ | $6.0\times10^4$ | | Wang et al. (2017) | Q | 80, 239 |
| VNKKGGMTAOGJCS-UHFFFAOYSA-N | $1.3\times10^1$ | | Wang et al. (2017) | Q | 80, 240 |
| MCM:C917OOH | $2.9\times10^6$ | | Wang et al. (2017) | Q | 80, 238 |
| $C_9H_{14}O_4$ | $1.2\times10^7$ | | Wang et al. (2017) | Q | 80, 239 |
| USPARXGTYFQAAO-UHFFFAOYSA-N | $1.2\times10^6$ | | Wang et al. (2017) | Q | 80, 240 |
| MCM:C91OOH | $1.7\times10^3$ | | Wang et al. (2017) | Q | 80, 238 |
| $C_9H_{18}O_3$ | $4.9\times10^2$ | | Wang et al. (2017) | Q | 80, 239 |
| LTHHCRGMHVQJHW-UHFFFAOYSA-N | $4.5\times10^2$ | | Wang et al. (2017) | Q | 80, 240 |
| MCM:C923OOH | $1.1\times10^4$ | 12000 | Wieser et al. (2023) | Q | 437 |
| $C_9H_{16}O_3$ | $4.1\times10^3$ | | Wang et al. (2017) | Q | 80, 238 |
| ZSCOJLRNNZZYOH-UHFFFAOYSA-N | $2.2\times10^3$ | | Wang et al. (2017) | Q | 80, 239 |
| | $2.9\times10^2$ | | Wang et al. (2017) | Q | 80, 240 |
| MCM:C928OOH | $8.0\times10^5$ | | Wang et al. (2017) | Q | 80, 238 |
| $C_9H_{16}O_4$ | $1.4\times10^6$ | | Wang et al. (2017) | Q | 80, 239 |
| NVMDAOHKHIQTQS-UHFFFAOYSA-N | $1.2\times10^4$ | | Wang et al. (2017) | Q | 80, 240 |
| MCM:C94OOH | $1.3\times10^6$ | | Wang et al. (2017) | Q | 80, 238 |
| $C_9H_{16}O_4$ | $9.3\times10^3$ | | Wang et al. (2017) | Q | 80, 239 |
| ZEMHBLLRPWJDGD-UHFFFAOYSA-N | $2.8\times10^2$ | | Wang et al. (2017) | Q | 80, 240 |
| MCM:C95OOH | $8.7\times10^8$ | | Wang et al. (2017) | Q | 80, 238 |
| $C_9H_{14}O_5$ | $2.3\times10^5$ | | Wang et al. (2017) | Q | 80, 239 |
| OKPNQXXXVOPUSS-UHFFFAOYSA-N | $5.4$ | | Wang et al. (2017) | Q | 80, 240 |





Table A3.6: Ketones (RCOR) (... continued)

| Substance<br>Formula<br>(Trivial Name)<br>[CAS Registry Number]<br>InChIKey | $H_s^{cp}$<br>(at $T^{\ominus}$)<br>$\left[\dfrac{\mathrm{mol}}{\mathrm{m^3\,Pa}}\right]$ | $\dfrac{\mathrm{d}\ln H_s^{cp}}{\mathrm{d}(1/T)}$<br><br>[K] | Reference | Type | Note |
|---|---|---|---|---|---|
| MCM:C96OOH<br>$C_9H_{16}O_3$<br>UOSOYGOFRNBVGT-UHFFFAOYSA-N | $4.4\times10^3$<br>$6.9\times10^3$<br>$4.8\times10^4$ | | Wang et al. (2017)<br>Wang et al. (2017)<br>Wang et al. (2017) | Q<br>Q<br>Q | 80, 238<br>80, 239<br>80, 240 |
| MCM:C9DC<br>$C_9H_{12}O_2$<br>MRLMBLCAGJIZHM-UHFFFAOYSA-N | $1.2\times10^2$<br>$2.3\times10^3$<br>$4.0\times10^3$ | | Wang et al. (2017)<br>Wang et al. (2017)<br>Wang et al. (2017) | Q<br>Q<br>Q | 80, 238<br>80, 239<br>80, 240 |
| MCM:C9DCCO<br>$C_9H_{10}O_3$<br>CNGBBXSWKSNDLL-UHFFFAOYSA-N | $9.6\times10^4$<br>$7.6\times10^5$<br>$2.4\times10^3$ | | Wang et al. (2017)<br>Wang et al. (2017)<br>Wang et al. (2017) | Q<br>Q<br>Q | 80, 238<br>80, 239<br>80, 240 |
| MCM:C9DCOOH<br>$C_9H_{12}O_4$<br>LVYPIBNFNZWAIN-UHFFFAOYSA-N | $1.1\times10^7$<br>$8.3\times10^7$<br>$2.8\times10^6$ | | Wang et al. (2017)<br>Wang et al. (2017)<br>Wang et al. (2017) | Q<br>Q<br>Q | 80, 238<br>80, 239<br>80, 240 |
| MCM:CO356C9<br>$C_9H_{14}O_3$<br>UAWDJSGDSYOWDW-UHFFFAOYSA-N | $1.0\times10^4$<br>$1.1\times10^3$<br>5.3 | | Wang et al. (2017)<br>Wang et al. (2017)<br>Wang et al. (2017) | Q<br>Q<br>Q | 80, 238<br>80, 239<br>80, 240 |
| MCM:CO36C9<br>$C_9H_{16}O_2$<br>NTESOQURXNMSRF-UHFFFAOYSA-N | $1.5\times10^1$<br>$1.0\times10^2$<br>$2.5\times10^1$ | | Wang et al. (2017)<br>Wang et al. (2017)<br>Wang et al. (2017) | Q<br>Q<br>Q | 80, 238<br>80, 239<br>80, 240 |
| MCM:IPRBQONE<br>$C_9H_{10}O_2$<br>XLTSBDOZTUSCMX-UHFFFAOYSA-N | 2.5<br>$2.4\times10^3$<br>8.7 | | Wang et al. (2017)<br>Wang et al. (2017)<br>Wang et al. (2017) | Q<br>Q<br>Q | 80, 238<br>80, 239<br>80, 240 |
| MCM:LIMKET<br>$C_9H_{14}O$<br>HOBBEYSRFFJETF-UHFFFAOYSA-N | $6.4\times10^{-1}$<br>$2.4\times10^{-1}$<br>$9.8\times10^{-1}$<br>3.2 | 8000 | Wieser et al. (2023)<br>Wang et al. (2017)<br>Wang et al. (2017)<br>Wang et al. (2017) | Q<br>Q<br>Q<br>Q | 437<br>80, 238<br>80, 239<br>80, 240 |
| MCM:METLQONE<br>$C_9H_{10}O_2$<br>FBVFNJVAQSWLOP-UHFFFAOYSA-N | 1.8<br>$3.7\times10^3$<br>$1.5\times10^1$ | | Wang et al. (2017)<br>Wang et al. (2017)<br>Wang et al. (2017) | Q<br>Q<br>Q | 80, 238<br>80, 239<br>80, 240 |
| MCM:NOPINAOOH<br>$C_9H_{14}O_3$<br>AGOGLNHOAIUZEO-UHFFFAOYSA-N | $1.4\times10^4$<br>$2.9\times10^4$<br>$8.3\times10^4$ | | Wang et al. (2017)<br>Wang et al. (2017)<br>Wang et al. (2017) | Q<br>Q<br>Q | 80, 238<br>80, 239<br>80, 240 |
| MCM:NOPINBCO<br>$C_9H_{12}O_2$<br>AQDJLLQBBRXMBZ-UHFFFAOYSA-N | $1.2\times10^2$<br>$2.2\times10^3$<br>$4.7\times10^2$ | | Wang et al. (2017)<br>Wang et al. (2017)<br>Wang et al. (2017) | Q<br>Q<br>Q | 80, 238<br>80, 239<br>80, 240 |
| MCM:NOPINBOOH<br>$C_9H_{14}O_3$<br>ICIAOEDGJQUKIV-UHFFFAOYSA-N | $1.4\times10^4$<br>$4.0\times10^4$<br>$9.3\times10^4$ | | Wang et al. (2017)<br>Wang et al. (2017)<br>Wang et al. (2017) | Q<br>Q<br>Q | 80, 238<br>80, 239<br>80, 240 |
| MCM:NOPINCOOH<br>$C_9H_{14}O_3$<br>YDBPVAMAGOPNKC-UHFFFAOYSA-N | $8.5\times10^3$<br>$1.3\times10^4$<br>$7.1\times10^4$ | | Wang et al. (2017)<br>Wang et al. (2017)<br>Wang et al. (2017) | Q<br>Q<br>Q | 80, 238<br>80, 239<br>80, 240 |



Table A3.6: Ketones (RCOR) (... continued)

| Substance Formula (Trivial Name) [CAS Registry Number] InChIKey | $H_s^{cp}$ (at $T^{\ominus}$) $\left[\dfrac{\mathrm{mol}}{\mathrm{m}^3\,\mathrm{Pa}}\right]$ | $\dfrac{\mathrm{d}\ln H_s^{cp}}{\mathrm{d}(1/T)}$ [K] | Reference | Type | Note |
|---|---|---|---|---|---|
| MCM:NOPINDCO $C_9H_{12}O_2$ QEZVNLZLSDOZRT-UHFFFAOYSA-N | $1.2\times10^2$ $5.0\times10^2$ $8.9\times10^2$ | | Wang et al. (2017) Wang et al. (2017) Wang et al. (2017) | Q Q Q | 80, 238 80, 239 80, 240 |
| MCM:NOPINDOOH $C_9H_{14}O_3$ MUTHAEIFCDWSQQ-UHFFFAOYSA-N | $1.4\times10^4$ $5.4\times10^2$ $1.4\times10^2$ | | Wang et al. (2017) Wang et al. (2017) Wang et al. (2017) | Q Q Q | 80, 238 80, 239 80, 240 |
| MCM:NOPINONE $C_9H_{14}O$ XZFDKWMYCUEKSS-UHFFFAOYSA-N | $1.8\times10^{-1}$ $1.0$ $1.4\times10^1$ | | Wang et al. (2017) Wang et al. (2017) Wang et al. (2017) | Q Q Q | 80, 238 80, 239 80, 240 |
| MCM:OETLQONE $C_9H_{10}O_2$ CMBANAGVBLPIPT-UHFFFAOYSA-N | $1.8$ $3.7\times10^3$ $8.1$ | | Wang et al. (2017) Wang et al. (2017) Wang et al. (2017) | Q Q Q | 80, 238 80, 239 80, 240 |
| MCM:PETLQONE $C_9H_{10}O_2$ JUIQOKRNPGGIPV-UHFFFAOYSA-N | $1.8$ $3.7\times10^3$ $1.3\times10^1$ | | Wang et al. (2017) Wang et al. (2017) Wang et al. (2017) | Q Q Q | 80, 238 80, 239 80, 240 |
| MCM:PPRBQONE $C_9H_{10}O_2$ NBHAZVWKRHTWRW-UHFFFAOYSA-N | $2.2$ $3.0\times10^3$ $1.4\times10^1$ | | Wang et al. (2017) Wang et al. (2017) Wang et al. (2017) | Q Q Q | 80, 238 80, 239 80, 240 |
| MCM:TM124QONE $C_9H_{10}O_2$ QIXDHVDGPXBRRD-UHFFFAOYSA-N | $1.4$ $4.1\times10^3$ $7.3$ | | Wang et al. (2017) Wang et al. (2017) Wang et al. (2017) | Q Q Q | 80, 238 80, 239 80, 240 |
| MCM:C1011CO $C_{10}H_{16}O_2$ HCTKOTXYXFZDNL-UHFFFAOYSA-N | $3.1\times10^1$ $3.3\times10^2$ $1.6\times10^3$ | | Wang et al. (2017) Wang et al. (2017) Wang et al. (2017) | Q Q Q | 80, 238 80, 239 80, 240 |
| MCM:C1011OOH $C_{10}H_{18}O_3$ SNMQVMGNQAVOQP-UHFFFAOYSA-N | $3.6\times10^3$ $4.3\times10^3$ $1.4\times10^3$ | | Wang et al. (2017) Wang et al. (2017) Wang et al. (2017) | Q Q Q | 80, 238 80, 239 80, 240 |
| MCM:C101OOH $C_{10}H_{20}O_3$ SFSDJTUNVGVIHA-UHFFFAOYSA-N | $1.4\times10^3$ $4.4\times10^2$ $3.3\times10^2$ | | Wang et al. (2017) Wang et al. (2017) Wang et al. (2017) | Q Q Q | 80, 238 80, 239 80, 240 |
| MCM:C104OOH $C_{10}H_{18}O_4$ NAKDXHBWPFMZPE-UHFFFAOYSA-N | $1.0\times10^6$ $6.9\times10^3$ $2.8\times10^2$ | | Wang et al. (2017) Wang et al. (2017) Wang et al. (2017) | Q Q Q | 80, 238 80, 239 80, 240 |
| MCM:C105OOH $C_{10}H_{16}O_5$ SMUSKFONQIUMCM-UHFFFAOYSA-N | $7.1\times10^8$ $1.7\times10^5$ $2.9$ | | Wang et al. (2017) Wang et al. (2017) Wang et al. (2017) | Q Q Q | 80, 238 80, 239 80, 240 |
| MCM:C923CO3H $C_{10}H_{16}O_4$ JSHGCXGTVADTAK-UHFFFAOYSA-N | $2.2\times10^4$ $4.7\times10^4$ $1.0\times10^3$ $3.1\times10^2$ | 15000 | Wieser et al. (2023) Wang et al. (2017) Wang et al. (2017) Wang et al. (2017) | Q Q Q Q | 437 80, 238 80, 239 80, 240 |



Table A3.6: Ketones (RCOR) (...continued)

| Substance<br>Formula<br>(Trivial Name)<br>[CAS Registry Number]<br>InChIKey | $H_s^{cp}$<br>(at $T^{\ominus}$)<br>$\left[\dfrac{\text{mol}}{\text{m}^3\,\text{Pa}}\right]$ | $\dfrac{\text{d}\ln H_s^{cp}}{\text{d}(1/T)}$<br><br>[K] | Reference | Type | Note |
|---|---|---|---|---|---|
| MCM:C928CO3H | $9.1\times10^6$ | | Wang et al. (2017) | Q | 80, 238 |
| $C_{10}H_{16}O_5$ | $1.5\times10^6$ | | Wang et al. (2017) | Q | 80, 239 |
| LCMHUZKVXFPENZ-UHFFFAOYSA-N | $1.7\times10^3$ | | Wang et al. (2017) | Q | 80, 240 |
| MCM:C9M2CO6OOH | $3.6\times10^6$ | | Wang et al. (2017) | Q | 80, 238 |
| $C_{10}H_{16}O_4$ | $1.9\times10^4$ | | Wang et al. (2017) | Q | 80, 239 |
| ACPAJMUJFSDSAX-UHFFFAOYSA-N | $9.6\times10^3$ | | Wang et al. (2017) | Q | 80, 240 |
| MCM:C9M3CO | $3.1\times10^4$ | | Wang et al. (2017) | Q | 80, 238 |
| $C_{10}H_{14}O_3$ | $3.5\times10^4$ | | Wang et al. (2017) | Q | 80, 239 |
| DYAPBQRZCBYWEA-UHFFFAOYSA-N | 9.3 | | Wang et al. (2017) | Q | 80, 240 |
| MCM:CO356C10 | $8.7\times10^3$ | | Wang et al. (2017) | Q | 80, 238 |
| $C_{10}H_{16}O_3$ | $9.1\times10^2$ | | Wang et al. (2017) | Q | 80, 239 |
| OKXFSYMNZFSQOL-UHFFFAOYSA-N | 4.7 | | Wang et al. (2017) | Q | 80, 240 |
| MCM:CO36C10 | $1.2\times10^1$ | | Wang et al. (2017) | Q | 80, 238 |
| $C_{10}H_{18}O_2$ | $8.0\times10^1$ | | Wang et al. (2017) | Q | 80, 239 |
| WYDYPTCJLWEFAY-UHFFFAOYSA-N | $2.6\times10^1$ | | Wang et al. (2017) | Q | 80, 240 |
| MCM:DEC3ONE | $1.8\times10^{-2}$ | | Wang et al. (2017) | Q | 80, 238 |
| $C_{10}H_{20}O$ | $2.6\times10^{-2}$ | | Wang et al. (2017) | Q | 80, 239 |
| XJLDYKIEURAVBW-UHFFFAOYSA-N | $1.0\times10^{-1}$ | | Wang et al. (2017) | Q | 80, 240 |
| MCM:PERPINONIC | $5.3\times10^4$ | | Wang et al. (2017) | Q | 80, 238 |
| $C_{10}H_{16}O_4$ | $3.0\times10^3$ | | Wang et al. (2017) | Q | 80, 239 |
| VFMYUIAYOIIENK-UHFFFAOYSA-N | $5.3\times10^2$ | | Wang et al. (2017) | Q | 80, 240 |
| MCM:C1011CO3H | $4.2\times10^4$ | | Wang et al. (2017) | Q | 80, 238 |
| $C_{11}H_{18}O_4$ | $2.5\times10^3$ | | Wang et al. (2017) | Q | 80, 239 |
| MDKPIQZZHUIYJQ-UHFFFAOYSA-N | $1.6\times10^2$ | | Wang et al. (2017) | Q | 80, 240 |
| MCM:C111OOH | $1.3\times10^3$ | | Wang et al. (2017) | Q | 80, 238 |
| $C_{11}H_{22}O_3$ | $3.6\times10^2$ | | Wang et al. (2017) | Q | 80, 239 |
| AMIJEMPORNLGBW-UHFFFAOYSA-N | $2.5\times10^2$ | | Wang et al. (2017) | Q | 80, 240 |
| MCM:C114OOH | $8.3\times10^5$ | | Wang et al. (2017) | Q | 80, 238 |
| $C_{11}H_{20}O_4$ | $5.4\times10^3$ | | Wang et al. (2017) | Q | 80, 239 |
| YQNGKLISCMALPY-UHFFFAOYSA-N | $4.6\times10^2$ | | Wang et al. (2017) | Q | 80, 240 |
| MCM:C115OOH | $6.3\times10^8$ | | Wang et al. (2017) | Q | 80, 238 |
| $C_{11}H_{18}O_5$ | $1.3\times10^5$ | | Wang et al. (2017) | Q | 80, 239 |
| VPBZYPGTLQIJKG-UHFFFAOYSA-N | 6.0 | | Wang et al. (2017) | Q | 80, 240 |
| MCM:CO356C11 | $7.3\times10^3$ | | Wang et al. (2017) | Q | 80, 238 |
| $C_{11}H_{18}O_3$ | $7.8\times10^2$ | | Wang et al. (2017) | Q | 80, 239 |
| YADKKYWEPSBFCL-UHFFFAOYSA-N | 4.9 | | Wang et al. (2017) | Q | 80, 240 |
| MCM:CO36C11 | $1.1\times10^1$ | | Wang et al. (2017) | Q | 80, 238 |
| $C_{11}H_{20}O_2$ | $6.5\times10^1$ | | Wang et al. (2017) | Q | 80, 239 |
| KWHQQLQVDJMZTG-UHFFFAOYSA-N | $2.3\times10^1$ | | Wang et al. (2017) | Q | 80, 240 |





Table A3.6: Ketones (RCOR) (...continued)

| Substance Formula (Trivial Name) [CAS Registry Number] InChIKey | $H_s^{cp}$ (at $T^{\ominus}$) $\left[\dfrac{\mathrm{mol}}{\mathrm{m^3\,Pa}}\right]$ | $\dfrac{\mathrm{d}\ln H_s^{cp}}{\mathrm{d}(1/T)}$ [K] | Reference | Type | Note |
|---|---|---|---|---|---|
| MCM:UDEC3ONE | $1.4\times10^{-2}$ | | Wang et al. (2017) | Q | 80, 238 |
| $C_{11}H_{22}O$ | $2.1\times10^{-2}$ | | Wang et al. (2017) | Q | 80, 239 |
| YNMZZHPSYMOGCI-UHFFFAOYSA-N | $1.0\times10^{-1}$ | | Wang et al. (2017) | Q | 80, 240 |
| MCM:C121OOH | $9.8\times10^2$ | | Wang et al. (2017) | Q | 80, 238 |
| $C_{12}H_{24}O_3$ | $2.8\times10^2$ | | Wang et al. (2017) | Q | 80, 239 |
| NWJGDRGQGOZUDJ-UHFFFAOYSA-N | $2.3\times10^2$ | | Wang et al. (2017) | Q | 80, 240 |
| MCM:C124OOH | $6.6\times10^5$ | | Wang et al. (2017) | Q | 80, 238 |
| $C_{12}H_{22}O_4$ | $4.8\times10^3$ | | Wang et al. (2017) | Q | 80, 239 |
| DCPRRPHGFZBUMW-UHFFFAOYSA-N | $9.3\times10^2$ | | Wang et al. (2017) | Q | 80, 240 |
| MCM:C125OOH | $4.9\times10^8$ | | Wang et al. (2017) | Q | 80, 238 |
| $C_{12}H_{20}O_5$ | $1.1\times10^5$ | | Wang et al. (2017) | Q | 80, 239 |
| FTFXETPWXOMYNK-UHFFFAOYSA-N | $2.1\times10^1$ | | Wang et al. (2017) | Q | 80, 240 |
| MCM:CO356C12 | $5.6\times10^3$ | | Wang et al. (2017) | Q | 80, 238 |
| $C_{12}H_{20}O_3$ | $7.3\times10^2$ | | Wang et al. (2017) | Q | 80, 239 |
| FQYVJQVZHPFSKC-UHFFFAOYSA-N | $3.9$ | | Wang et al. (2017) | Q | 80, 240 |
| MCM:CO36C12 | $8.5$ | | Wang et al. (2017) | Q | 80, 238 |
| $C_{12}H_{22}O_2$ | $5.8\times10^1$ | | Wang et al. (2017) | Q | 80, 239 |
| RUTQSYKISNASLK-UHFFFAOYSA-N | $2.1\times10^1$ | | Wang et al. (2017) | Q | 80, 240 |
| MCM:DDEC3ONE | $1.1\times10^{-2}$ | | Wang et al. (2017) | Q | 80, 238 |
| $C_{12}H_{24}O$ | $1.7\times10^{-2}$ | | Wang et al. (2017) | Q | 80, 239 |
| PERIHWAPLOBAJM-UHFFFAOYSA-N | $9.6\times10^{-2}$ | | Wang et al. (2017) | Q | 80, 240 |
| MCM:C131OOH | $1.5\times10^6$ | | Wang et al. (2017) | Q | 80, 238 |
| $C_{13}H_{22}O_4$ | $2.0\times10^6$ | | Wang et al. (2017) | Q | 80, 239 |
| HPZPYFVEAXAGHY-UHFFFAOYSA-N | $1.4\times10^5$ | | Wang et al. (2017) | Q | 80, 240 |
| MCM:BCKET | $2.3\times10^{-1}$ | | Wang et al. (2017) | Q | 80, 238 |
| $C_{14}H_{22}O$ | $2.6$ | | Wang et al. (2017) | Q | 80, 239 |
| MBZBBVTYLUNZPJ-UHFFFAOYSA-N | $2.9\times10^1$ | | Wang et al. (2017) | Q | 80, 240 |
| MCM:C131CO3H | $1.7\times10^7$ | | Wang et al. (2017) | Q | 80, 238 |
| $C_{14}H_{22}O_5$ | $2.1\times10^6$ | | Wang et al. (2017) | Q | 80, 239 |
| LLSRBTGTUZNRQT-UHFFFAOYSA-N | $8.3\times10^5$ | | Wang et al. (2017) | Q | 80, 240 |
| MCM:C141OOH | $3.8\times10^3$ | | Wang et al. (2017) | Q | 80, 238 |
| $C_{14}H_{24}O_3$ | $4.9\times10^3$ | | Wang et al. (2017) | Q | 80, 239 |
| HKWHMEKBBPOBSQ-UHFFFAOYSA-N | $7.8\times10^2$ | | Wang et al. (2017) | Q | 80, 240 |
| MCM:C141CO3H | $4.6\times10^4$ | | Wang et al. (2017) | Q | 80, 238 |
| $C_{15}H_{24}O_4$ | $1.8\times10^3$ | | Wang et al. (2017) | Q | 80, 239 |
| ADRMPTUALHKSOU-UHFFFAOYSA-N | $1.4\times10^3$ | | Wang et al. (2017) | Q | 80, 240 |
| MCM:C619CO | $5.3\times10^4$ | | Wang et al. (2017) | Q | 80, 238 |
| $C_6H_6O_3$ | $1.1\times10^6$ | | Wang et al. (2017) | Q | 80, 239 |
| NGHVTTBOPYXCFB-UHFFFAOYSA-N | $1.1\times10^5$ | | Wang et al. (2017) | Q | 80, 240 |





Table A3.6: Ketones (RCOR) (. . . continued)

| Substance Formula (Trivial Name) [CAS Registry Number] InChIKey | $H_s^{cp}$ (at $T^{\ominus}$) $\left[\dfrac{\mathrm{mol}}{\mathrm{m}^3\,\mathrm{Pa}}\right]$ | $\dfrac{\mathrm{d}\ln H_s^{cp}}{\mathrm{d}(1/T)}$ [K] | Reference | Type | Note |
|---|---|---|---|---|---|
| MCM:CY6TRION $C_6H_6O_3$ GKSCYYCYSPXQFY-UHFFFAOYSA-N | $5.3\times10^4$ $4.6\times10^6$ $1.9\times10^5$ | | Wang et al. (2017) Wang et al. (2017) Wang et al. (2017) | Q Q Q | 80, 238 80, 239 80, 240 |
| MCM:PHCOMEOOH $C_8H_8O_3$ BXVCLOREWFXMNM-UHFFFAOYSA-N | $1.6\times10^5$ $1.4\times10^3$ $3.7\times10^2$ | | Wang et al. (2017) Wang et al. (2017) Wang et al. (2017) | Q Q Q | 80, 238 80, 239 80, 240 |
| MCM:MPHCOMEOOH $C_9H_{10}O_3$ NJDRUBYBGCJLGU-UHFFFAOYSA-N | $9.1\times10^4$ $1.4\times10^3$ $3.1\times10^2$ | | Wang et al. (2017) Wang et al. (2017) Wang et al. (2017) | Q Q Q | 80, 238 80, 239 80, 240 |
| MCM:PHCOCOCOOH $C_9H_8O_4$ XMJMKOZGMVOAIC-UHFFFAOYSA-N | $9.1\times10^7$ $4.7\times10^4$ $5.6\times10^2$ | | Wang et al. (2017) Wang et al. (2017) Wang et al. (2017) | Q Q Q | 80, 238 80, 239 80, 240 |
| MCM:PHCOCOME $C_9H_8O_2$ BVQVLAIMHVDZEL-UHFFFAOYSA-N | $1.3\times10^3$ $9.1\times10^1$ $6.9$ | | Wang et al. (2017) Wang et al. (2017) Wang et al. (2017) | Q Q Q | 80, 238 80, 239 80, 240 |
| MCM:PHCOETOOH $C_9H_{10}O_3$ RYQHVKZCKUBIFL-UHFFFAOYSA-N | $1.5\times10^5$ $6.0\times10^2$ $3.7\times10^2$ | | Wang et al. (2017) Wang et al. (2017) Wang et al. (2017) | Q Q Q | 80, 238 80, 239 80, 240 |
| MCM:DMPHCOME $C_{10}H_{12}O$ BKIHFZLJJUNKMZ-UHFFFAOYSA-N | $7.4\times10^{-1}$ $1.5$ $5.4$ | | Wang et al. (2017) Wang et al. (2017) Wang et al. (2017) | Q Q Q | 80, 238 80, 239 80, 240 |
| MCM:DMPHCOMOOH $C_{10}H_{12}O_3$ SEDWELDPOZYWLL-UHFFFAOYSA-N | $5.4\times10^4$ $8.5\times10^2$ $3.0\times10^2$ | | Wang et al. (2017) Wang et al. (2017) Wang et al. (2017) | Q Q Q | 80, 238 80, 239 80, 240 |
| MCM:EMPHCOME $C_{11}H_{14}O$ CZNQKVDFQIGBEW-UHFFFAOYSA-N | $6.5\times10^{-1}$ $1.2$ $5.0$ | | Wang et al. (2017) Wang et al. (2017) Wang et al. (2017) | Q Q Q | 80, 238 80, 239 80, 240 |
| MCM:EMPHCOMOOH $C_{11}H_{14}O_3$ OJNDCDHAKIGFTB-UHFFFAOYSA-N | $5.0\times10^4$ $5.0\times10^2$ $2.3\times10^2$ | | Wang et al. (2017) Wang et al. (2017) Wang et al. (2017) | Q Q Q | 80, 238 80, 239 80, 240 |
| MCM:BIACETOH $C_4H_6O_3$ NXIVQSJQXMAXJR-UHFFFAOYSA-N | $5.1\times10^3$ $4.3\times10^3$ $3.7$ | | Wang et al. (2017) Wang et al. (2017) Wang et al. (2017) | Q Q Q | 80, 238 80, 239 80, 240 |
| MCM:CO2H3CO3H $C_4H_6O_5$ WNDCXKJVOGKFRM-UHFFFAOYSA-N | $2.0\times10^6$ $1.7\times10^4$ $1.4\times10^2$ | | Wang et al. (2017) Wang et al. (2017) Wang et al. (2017) | Q Q Q | 80, 238 80, 239 80, 240 |
| MCM:H13CO2CO3H $C_4H_6O_6$ RXVYYLAFXUAZCR-UHFFFAOYSA-N | $1.2\times10^9$ $4.0\times10^5$ $1.7\times10^5$ | | Wang et al. (2017) Wang et al. (2017) Wang et al. (2017) | Q Q Q | 80, 238 80, 239 80, 240 |





Table A3.6: Ketones (RCOR) (...continued)

| Substance Formula (Trivial Name) [CAS Registry Number] InChIKey | $H_s^{cp}$ (at $T^{\ominus}$) $\left[\dfrac{\text{mol}}{\text{m}^3\,\text{Pa}}\right]$ | $\dfrac{\text{d}\ln H_s^{cp}}{\text{d}(1/T)}$ [K] | Reference | Type | Note |
|---|---|---|---|---|---|
| MCM:H14CO23C4 $C_4H_6O_4$ GJCZUCLKDGABDS-UHFFFAOYSA-N | $2.1\times10^6$ $6.5\times10^5$ $1.0\times10^6$ $1.6\times10^3$ | 17000 | Wieser et al. (2023) Wang et al. (2017) Wang et al. (2017) Wang et al. (2017) | Q Q Q Q | 437 80, 238 80, 239 80, 240 |
| MCM:H1C23C4OOH $C_4H_6O_5$ FQXHRCVOGVCTSR-UHFFFAOYSA-N | $3.9\times10^8$ $6.0\times10^7$ $3.7\times10^3$ | | Wang et al. (2017) Wang et al. (2017) Wang et al. (2017) | Q Q Q | 80, 238 80, 239 80, 240 |
| MCM:HMVKAOOH $C_4H_8O_4$ XTELZFGIGJBUGR-UHFFFAOYSA-N | $8.3\times10^5$ $7.8\times10^4$ $1.3\times10^4$ | | Wang et al. (2017) Wang et al. (2017) Wang et al. (2017) | Q Q Q | 80, 238 80, 239 80, 240 |
| MCM:HMVKBOOH $C_4H_8O_4$ NGLQGLCPXKPDMS-UHFFFAOYSA-N | $1.6\times10^7$ $2.0\times10^5$ $1.2\times10^3$ | | Wang et al. (2017) Wang et al. (2017) Wang et al. (2017) | Q Q Q | 80, 238 80, 239 80, 240 |
| MCM:HO12CO3C4 $C_4H_8O_3$ SEYLPRWNVFCVRQ-UHFFFAOYSA-N | $9.8\times10^3$ $3.7\times10^4$ $1.0\times10^3$ | | Wang et al. (2017) Wang et al. (2017) Wang et al. (2017) | Q Q Q | 80, 238 80, 239 80, 240 |
| MCM:HO14CO2C4 $C_4H_8O_3$ XBJODPUPYBBDEM-UHFFFAOYSA-N | $1.9\times10^4$ $9.8\times10^5$ $1.0\times10^4$ | | Wang et al. (2017) Wang et al. (2017) Wang et al. (2017) | Q Q Q | 80, 238 80, 239 80, 240 |
| MCM:HOCO3C4OOH $C_4H_8O_4$ GQXQZBROPOTXJQ-UHFFFAOYSA-N | $1.0\times10^7$ $1.9\times10^6$ $1.0\times10^5$ | | Wang et al. (2017) Wang et al. (2017) Wang et al. (2017) | Q Q Q | 80, 238 80, 239 80, 240 |
| MCM:MEKAOH $C_4H_8O_2$ LVSQXDHWDCMMRJ-UHFFFAOYSA-N | $1.5\times10^2$ $1.2\times10^3$ $2.0\times10^2$ | | Wang et al. (2017) Wang et al. (2017) Wang et al. (2017) | Q Q Q | 80, 238 80, 239 80, 240 |
| MCM:MEKCOH $C_4H_8O_2$ GFAZHVHNLUBROE-UHFFFAOYSA-N | 7.8 $7.8\times10^1$ 8.3 | | Wang et al. (2017) Wang et al. (2017) Wang et al. (2017) | Q Q Q | 80, 238 80, 239 80, 240 |
| MCM:MVKOHAOH $C_4H_8O_4$ UQPHVQVXLPRNCX-UHFFFAOYSA-N | $6.2\times10^6$ $3.7\times10^6$ $8.9\times10^4$ | | Wang et al. (2017) Wang et al. (2017) Wang et al. (2017) | Q Q Q | 80, 238 80, 239 80, 240 |
| MCM:MVKOHAOOH $C_4H_8O_5$ MOVXURHCNUWOQO-UHFFFAOYSA-N | $4.6\times10^8$ $2.2\times10^7$ $1.8\times10^6$ | | Wang et al. (2017) Wang et al. (2017) Wang et al. (2017) | Q Q Q | 80, 238 80, 239 80, 240 |
| MCM:MVKOHBOOH $C_4H_8O_5$ WSMWIMXDHMPHIN-UHFFFAOYSA-N | $2.0\times10^9$ $2.7\times10^8$ $5.8\times10^5$ | | Wang et al. (2017) Wang et al. (2017) Wang et al. (2017) | Q Q Q | 80, 238 80, 239 80, 240 |
| MCM:MVKOH $C_4H_6O_2$ LHBQGXZUVXFJRH-UHFFFAOYSA-N | $2.1\times10^1$ $2.3\times10^2$ 4.9 | | Wang et al. (2017) Wang et al. (2017) Wang et al. (2017) | Q Q Q | 80, 238 80, 239 80, 240 |



Table A3.6: Ketones (RCOR) (...continued)

| Substance<br>Formula<br>(Trivial Name)<br>[CAS Registry Number]<br>InChIKey | $H_s^{cp}$<br>(at $T^\ominus$)<br><br>$\left[\dfrac{\text{mol}}{\text{m}^3\,\text{Pa}}\right]$ | $\dfrac{\text{d}\ln H_s^{cp}}{\text{d}(1/T)}$<br><br>[K] | Reference | Type | Note |
|---|---|---|---|---|---|
| MCM:C517OH<br>$C_5H_{10}O_3$<br>NQMZUMSFJBKHAU-UHFFFAOYSA-N | $3.8\times10^5$<br>$1.6\times10^6$<br>$3.6\times10^3$ | | Wang et al. (2017)<br>Wang et al. (2017)<br>Wang et al. (2017) | Q<br>Q<br>Q | 80, 238<br>80, 239<br>80, 240 |
| MCM:C517OOH<br>$C_5H_{10}O_4$<br>OAFCGFSFCVGKSH-UHFFFAOYSA-N | $1.1\times10^6$<br>$1.0\times10^7$<br>$4.9\times10^5$<br>$1.1\times10^4$ | 13000 | Wieser et al. (2023)<br>Wang et al. (2017)<br>Wang et al. (2017)<br>Wang et al. (2017) | Q<br>Q<br>Q<br>Q | 437<br>80, 238<br>80, 239<br>80, 240 |
| MCM:C519OOH<br>$C_5H_{10}O_4$<br>RDSFTYTWLRHJGZ-UHFFFAOYSA-N | $1.7\times10^5$<br>$1.0\times10^7$<br>$4.0\times10^5$<br>$8.0\times10^4$ | 13000 | Wieser et al. (2023)<br>Wang et al. (2017)<br>Wang et al. (2017)<br>Wang et al. (2017) | Q<br>Q<br>Q<br>Q | 437<br>80, 238<br>80, 239<br>80, 240 |
| MCM:C51OH2CO<br>$C_5H_{10}O_2$<br>WOVLKKLXYZJMSN-UHFFFAOYSA-N | $6.0$<br>$4.2\times10^1$<br>$9.6$ | | Wang et al. (2017)<br>Wang et al. (2017)<br>Wang et al. (2017) | Q<br>Q<br>Q | 80, 238<br>80, 239<br>80, 240 |
| MCM:C51OH<br>$C_5H_{10}O_3$<br>NJUQBPDKVORANW-UHFFFAOYSA-N | $1.5\times10^5$<br>$2.1\times10^5$<br>$4.9\times10^3$ | | Wang et al. (2017)<br>Wang et al. (2017)<br>Wang et al. (2017) | Q<br>Q<br>Q | 80, 238<br>80, 239<br>80, 240 |
| MCM:C51OOH<br>$C_5H_{10}O_4$<br>IZBMMNKBXWVQLB-UHFFFAOYSA-N | $1.3\times10^7$<br>$2.1\times10^6$<br>$1.6\times10^4$ | | Wang et al. (2017)<br>Wang et al. (2017)<br>Wang et al. (2017) | Q<br>Q<br>Q | 80, 238<br>80, 239<br>80, 240 |
| MCM:C521OH<br>$C_5H_8O_4$<br>UOGQGBWLBZOULD-UHFFFAOYSA-N | $1.2\times10^7$<br>$4.8\times10^7$<br>$9.6\times10^3$ | | Wang et al. (2017)<br>Wang et al. (2017)<br>Wang et al. (2017) | Q<br>Q<br>Q | 80, 238<br>80, 239<br>80, 240 |
| MCM:C521OOH<br>$C_5H_8O_5$<br>INSNYERMNWOCIO-UHFFFAOYSA-N | $6.5\times10^9$<br>$1.9\times10^7$<br>$2.7\times10^4$ | | Wang et al. (2017)<br>Wang et al. (2017)<br>Wang et al. (2017) | Q<br>Q<br>Q | 80, 238<br>80, 239<br>80, 240 |
| MCM:C524CO<br>$C_5H_8O_3$<br>JVFYSLPFISVGRS-UHFFFAOYSA-N | $3.5\times10^4$<br>$1.9\times10^5$<br>$1.1\times10^4$ | | Wang et al. (2017)<br>Wang et al. (2017)<br>Wang et al. (2017) | Q<br>Q<br>Q | 80, 238<br>80, 239<br>80, 240 |
| MCM:C525OOH<br>$C_5H_{10}O_6$<br>KYXSOONCNSTMJK-UHFFFAOYSA-N | $3.8\times10^{12}$<br>$2.0\times10^{11}$<br>$3.1\times10^6$ | | Wang et al. (2017)<br>Wang et al. (2017)<br>Wang et al. (2017) | Q<br>Q<br>Q | 80, 238<br>80, 239<br>80, 240 |
| MCM:C53OH<br>$C_5H_{10}O_3$<br>PLVCXNLHUHTLCR-UHFFFAOYSA-N | $1.8\times10^4$<br>$1.1\times10^6$<br>$6.0\times10^3$ | | Wang et al. (2017)<br>Wang et al. (2017)<br>Wang et al. (2017) | Q<br>Q<br>Q | 80, 238<br>80, 239<br>80, 240 |
| MCM:C53OOH<br>$C_5H_{10}O_4$<br>VCGFNBZCPLQTJR-UHFFFAOYSA-N | $1.0\times10^7$<br>$1.8\times10^6$<br>$6.2\times10^4$ | | Wang et al. (2017)<br>Wang et al. (2017)<br>Wang et al. (2017) | Q<br>Q<br>Q | 80, 238<br>80, 239<br>80, 240 |
| MCM:C55OOH<br>$C_5H_{10}O_4$<br>IEXMEMALQGQSKE-UHFFFAOYSA-N | $3.2\times10^5$<br>$1.3\times10^6$<br>$2.8\times10^3$ | | Wang et al. (2017)<br>Wang et al. (2017)<br>Wang et al. (2017) | Q<br>Q<br>Q | 80, 238<br>80, 239<br>80, 240 |





Table A3.6: Ketones (RCOR) (...continued)

| Substance Formula (Trivial Name) [CAS Registry Number] InChIKey | $H_s^{cp}$ (at $T^{\ominus}$) $\left[\dfrac{\text{mol}}{\text{m}^3\,\text{Pa}}\right]$ | $\dfrac{\text{d}\ln H_s^{cp}}{\text{d}(1/T)}$ [K] | Reference | Type | Note |
|---|---|---|---|---|---|
| MCM:C59OOH $C_5H_{10}O_5$ JWNCLQLBUYGJMN-UHFFFAOYSA-N | $1.1\times10^9$ $1.6\times10^8$ $3.0\times10^4$ | | Wang et al. (2017) Wang et al. (2017) Wang et al. (2017) | Q Q Q | 80, 238 80, 239 80, 240 |
| MCM:C5CO243OH $C_5H_8O_3$ SLKPFOVYRYRNBA-UHFFFAOYSA-N | $1.1\times10^3$ $1.1\times10^4$ 9.8 | | Wang et al. (2017) Wang et al. (2017) Wang et al. (2017) | Q Q Q | 80, 238 80, 239 80, 240 |
| MCM:C5OHCO4OOH $C_5H_{10}O_4$ DSLVXKSJLFFUGE-UHFFFAOYSA-N | $4.7\times10^5$ $1.6\times10^6$ $4.1\times10^4$ | | Wang et al. (2017) Wang et al. (2017) Wang et al. (2017) | Q Q Q | 80, 238 80, 239 80, 240 |
| MCM:CO2C5OH $C_5H_{10}O_2$ JSHPTIGHEWEXRW-UHFFFAOYSA-N | $3.6\times10^3$ $1.2\times10^2$ $2.6\times10^3$ $1.3\times10^3$ | 9800 | Wieser et al. (2023) Wang et al. (2017) Wang et al. (2017) Wang et al. (2017) | Q Q Q Q | 437 80, 238 80, 239 80, 240 |
| MCM:CO2H3MCO3H $C_5H_8O_5$ LZACDMRRAOGIBF-UHFFFAOYSA-N | $1.1\times10^6$ $3.2\times10^3$ $3.5\times10^1$ | | Wang et al. (2017) Wang et al. (2017) Wang et al. (2017) | Q Q Q | 80, 238 80, 239 80, 240 |
| MCM:CO3H4CO3H $C_5H_8O_5$ LAOPLCWFGWEKGR-UHFFFAOYSA-N | $1.6\times10^6$ $1.0\times10^4$ $1.8\times10^2$ | | Wang et al. (2017) Wang et al. (2017) Wang et al. (2017) | Q Q Q | 80, 238 80, 239 80, 240 |
| MCM:DIEKAOH $C_5H_{10}O_2$ QMXCHEVUAIPIRM-UHFFFAOYSA-N | 7.3 $6.8\times10^1$ 7.6 | | Wang et al. (2017) Wang et al. (2017) Wang et al. (2017) | Q Q Q | 80, 238 80, 239 80, 240 |
| MCM:H3C2C4CO3H $C_5H_8O_5$ NYZNEBYYPQANOB-UHFFFAOYSA-N | $6.9\times10^6$ $3.0\times10^6$ $6.3\times10^2$ | | Wang et al. (2017) Wang et al. (2017) Wang et al. (2017) | Q Q Q | 80, 238 80, 239 80, 240 |
| MCM:HC23C4CO3H $C_5H_6O_6$ UQNPRLJBDVQJQC-UHFFFAOYSA-N | $4.5\times10^9$ $1.1\times10^8$ $2.0\times10^3$ | | Wang et al. (2017) Wang et al. (2017) Wang et al. (2017) | Q Q Q | 80, 238 80, 239 80, 240 |
| MCM:HCOC5 $C_5H_8O_2$ LSMLKPXBSFFBNR-UHFFFAOYSA-N | $1.4\times10^1$ $1.4\times10^2$ 3.4 | | Wang et al. (2017) Wang et al. (2017) Wang et al. (2017) | Q Q Q | 80, 238 80, 239 80, 240 |
| MCM:HMVKBCO3H $C_5H_8O_5$ VFNFGUWSZFHBRW-UHFFFAOYSA-N | $1.4\times10^8$ $5.8\times10^6$ $2.1\times10^3$ | | Wang et al. (2017) Wang et al. (2017) Wang et al. (2017) | Q Q Q | 80, 238 80, 239 80, 240 |
| MCM:HO13CO4C5 $C_5H_{10}O_3$ CGPIPFJOMJUWGK-UHFFFAOYSA-N | $1.8\times10^4$ $2.6\times10^5$ $3.6\times10^3$ | | Wang et al. (2017) Wang et al. (2017) Wang et al. (2017) | Q Q Q | 80, 238 80, 239 80, 240 |
| MCM:HO14CO2C5 $C_5H_{10}O_3$ XLDKDOKPMNKVMP-UHFFFAOYSA-N | $1.8\times10^4$ $1.5\times10^6$ $4.2\times10^3$ | | Wang et al. (2017) Wang et al. (2017) Wang et al. (2017) | Q Q Q | 80, 238 80, 239 80, 240 |



Table A3.6: Ketones (RCOR) (. . . continued)

| Substance<br>Formula<br>(Trivial Name)<br>[CAS Registry Number]<br>InChIKey | $H_s^{cp}$<br>(at $T^\ominus$)<br>$\left[\dfrac{\mathrm{mol}}{\mathrm{m^3\,Pa}}\right]$ | $\dfrac{\mathrm{d}\ln H_s^{cp}}{\mathrm{d}(1/T)}$<br><br>[K] | Reference | Type | Note |
|---|---|---|---|---|---|
| MCM:HO14CO3C5<br>$C_5H_{10}O_3$<br>YMIRTILUDYUWGY-UHFFFAOYSA-N | $1.8\times10^4$<br>$1.6\times10^6$<br>$4.5\times10^3$ | | Wang et al. (2017)<br>Wang et al. (2017)<br>Wang et al. (2017) | Q<br>Q<br>Q | 80, 238<br>80, 239<br>80, 240 |
| MCM:HO1CO24C5<br>$C_5H_8O_3$<br>VBUBLWCIADYRMB-UHFFFAOYSA-N | $4.1\times10^3$<br>$3.2\times10^4$<br>$4.8\times10^2$ | | Wang et al. (2017)<br>Wang et al. (2017)<br>Wang et al. (2017) | Q<br>Q<br>Q | 80, 238<br>80, 239<br>80, 240 |
| MCM:HO1CO34C5<br>$C_5H_8O_3$<br>LKLKMCUVBQDYFU-UHFFFAOYSA-N | $7.8\times10^4$<br>$4.7\times10^4$<br>$1.1\times10^2$ | | Wang et al. (2017)<br>Wang et al. (2017)<br>Wang et al. (2017) | Q<br>Q<br>Q | 80, 238<br>80, 239<br>80, 240 |
| MCM:HO1CO3C5<br>$C_5H_{10}O_2$<br>TYXULUBCBKMSSK-UHFFFAOYSA-N | $1.2\times10^2$<br>$4.6\times10^2$<br>$6.2\times10^1$ | | Wang et al. (2017)<br>Wang et al. (2017)<br>Wang et al. (2017) | Q<br>Q<br>Q | 80, 238<br>80, 239<br>80, 240 |
| MCM:HO2CO4C5<br>$C_5H_{10}O_2$<br>PCYZZYAEGNVNMH-UHFFFAOYSA-N | $1.4\times10^2$<br>$1.3\times10^3$<br>$9.1\times10^1$ | | Wang et al. (2017)<br>Wang et al. (2017)<br>Wang et al. (2017) | Q<br>Q<br>Q | 80, 238<br>80, 239<br>80, 240 |
| MCM:HOCO3C5OOH<br>$C_5H_{10}O_4$<br>ZLPJDXTYGYNOQK-UHFFFAOYSA-N | $1.0\times10^7$<br>$1.3\times10^6$<br>$4.5\times10^4$ | | Wang et al. (2017)<br>Wang et al. (2017)<br>Wang et al. (2017) | Q<br>Q<br>Q | 80, 238<br>80, 239<br>80, 240 |
| MCM:HOCO4C5OOH<br>$C_5H_{10}O_4$<br>MYGJVQXASPJPCS-UHFFFAOYSA-N | $1.0\times10^7$<br>$1.9\times10^6$<br>$3.6\times10^4$ | | Wang et al. (2017)<br>Wang et al. (2017)<br>Wang et al. (2017) | Q<br>Q<br>Q | 80, 238<br>80, 239<br>80, 240 |
| MCM:MBOACO<br>$C_5H_{10}O_3$<br>FEIUXLCYYBVUGD-UHFFFAOYSA-N | $2.8\times10^3$<br>$9.8\times10^2$<br>$6.8\times10^2$ | | Wang et al. (2017)<br>Wang et al. (2017)<br>Wang et al. (2017) | Q<br>Q<br>Q | 80, 238<br>80, 239<br>80, 240 |
| MCM:ME3CO2BUOL<br>$C_5H_{10}O_2$<br>NBEGXSQMVJTIAR-UHFFFAOYSA-N | $7.3$<br>$4.6\times10^1$<br>$6.8$ | | Wang et al. (2017)<br>Wang et al. (2017)<br>Wang et al. (2017) | Q<br>Q<br>Q | 80, 238<br>80, 239<br>80, 240 |
| MCM:MIPKAOH<br>$C_5H_{10}O_2$<br>BNDRWEVUODOUDW-UHFFFAOYSA-N | $4.5$<br>$4.4\times10^1$<br>$7.6$ | | Wang et al. (2017)<br>Wang et al. (2017)<br>Wang et al. (2017) | Q<br>Q<br>Q | 80, 238<br>80, 239<br>80, 240 |
| MCM:MIPKBOH<br>$C_5H_{10}O_2$<br>VVSRECWZBBJOTG-UHFFFAOYSA-N | $1.4\times10^2$<br>$6.3\times10^2$<br>$6.3\times10^1$ | | Wang et al. (2017)<br>Wang et al. (2017)<br>Wang et al. (2017) | Q<br>Q<br>Q | 80, 238<br>80, 239<br>80, 240 |
| MCM:MPRKAOH<br>$C_5H_{10}O_2$<br>HDKKRASBPHFULQ-UHFFFAOYSA-N | $7.3$<br>$8.5\times10^1$<br>$4.9$ | | Wang et al. (2017)<br>Wang et al. (2017)<br>Wang et al. (2017) | Q<br>Q<br>Q | 80, 238<br>80, 239<br>80, 240 |
| MCM:BZOBIPEROH<br>$C_6H_6O_4$<br>OGRXSIBGKCRBOL-UHFFFAOYSA-N | $6.2\times10^6$<br>$3.6\times10^3$<br>$1.7\times10^7$ | | Wang et al. (2017)<br>Wang et al. (2017)<br>Wang et al. (2017) | Q<br>Q<br>Q | 80, 238<br>80, 239<br>80, 240 |



Table A3.6: Ketones (RCOR) (. . . continued)

| Substance Formula (Trivial Name) [CAS Registry Number] InChIKey | $H_s^{cp}$ (at $T^\ominus$) $\left[\dfrac{\text{mol}}{\text{m}^3\,\text{Pa}}\right]$ | $\dfrac{\text{d}\ln H_s^{cp}}{\text{d}(1/T)}$ [K] | Reference | Type | Note |
|---|---|---|---|---|---|
| MCM:C4COMEOH $C_6H_{12}O_2$ FDJJNIXWMAWMBP-UHFFFAOYSA-N | 5.0 $2.7\times10^1$ 8.0 | | Wang et al. (2017) Wang et al. (2017) Wang et al. (2017) | Q Q Q | 80, 238 80, 239 80, 240 |
| MCM:C4COMOH3OH $C_6H_{12}O_3$ NCEURKGQTPCVEC-UHFFFAOYSA-N | $1.6\times10^4$ $7.6\times10^5$ $1.7\times10^3$ | | Wang et al. (2017) Wang et al. (2017) Wang et al. (2017) | Q Q Q | 80, 238 80, 239 80, 240 |
| MCM:C4COMOHOOH $C_6H_{12}O_4$ JZGTXIGKACACHL-UHFFFAOYSA-N | $4.3\times10^5$ $8.9\times10^5$ $3.5\times10^4$ | | Wang et al. (2017) Wang et al. (2017) Wang et al. (2017) | Q Q Q | 80, 238 80, 239 80, 240 |
| MCM:C4MOHOCO3H $C_6H_{10}O_5$ BJXKNHIKUQDPQL-UHFFFAOYSA-N | $1.4\times10^6$ $5.4\times10^3$ $8.0\times10^1$ | | Wang et al. (2017) Wang et al. (2017) Wang et al. (2017) | Q Q Q | 80, 238 80, 239 80, 240 |
| MCM:C517CO3H $C_6H_{10}O_5$ HTNOTPIWPYZOLN-UHFFFAOYSA-N | $1.2\times10^8$ $1.1\times10^7$ $7.8\times10^3$ | | Wang et al. (2017) Wang et al. (2017) Wang et al. (2017) | Q Q Q | 80, 238 80, 239 80, 240 |
| MCM:C519CO3H $C_6H_{10}O_5$ OBKQTPZEBCGGJT-UHFFFAOYSA-N | $1.2\times10^8$ $1.6\times10^7$ $5.5\times10^3$ | | Wang et al. (2017) Wang et al. (2017) Wang et al. (2017) | Q Q Q | 80, 238 80, 239 80, 240 |
| MCM:C5COHOCO3H $C_6H_8O_6$ BPYCYKKOURHNJX-UHFFFAOYSA-N | $5.9\times10^8$ $2.3\times10^5$ $8.1\times10^1$ | | Wang et al. (2017) Wang et al. (2017) Wang et al. (2017) | Q Q Q | 80, 238 80, 239 80, 240 |
| MCM:C5O5OHCO3H $C_6H_{10}O_5$ GCQMJNYHSPBFDA-UHFFFAOYSA-N | $1.3\times10^6$ $4.8\times10^3$ $1.1\times10^2$ | | Wang et al. (2017) Wang et al. (2017) Wang et al. (2017) | Q Q Q | 80, 238 80, 239 80, 240 |
| MCM:C610OH $C_6H_{12}O_3$ YFEDHAOQJOZUIL-UHFFFAOYSA-N | $1.6\times10^5$ $2.4\times10^5$ $2.9\times10^3$ | | Wang et al. (2017) Wang et al. (2017) Wang et al. (2017) | Q Q Q | 80, 238 80, 239 80, 240 |
| MCM:C610OOH $C_6H_{12}O_4$ VBEFRWQFSDXQDV-UHFFFAOYSA-N | $1.2\times10^7$ $2.0\times10^6$ $6.5\times10^3$ | | Wang et al. (2017) Wang et al. (2017) Wang et al. (2017) | Q Q Q | 80, 238 80, 239 80, 240 |
| MCM:C611OH $C_6H_{10}O_3$ PYRMISJRKRVNPG-UHFFFAOYSA-N | $5.9\times10^2$ $3.0\times10^3$ 3.5 | | Wang et al. (2017) Wang et al. (2017) Wang et al. (2017) | Q Q Q | 80, 238 80, 239 80, 240 |
| MCM:C612OH $C_6H_{12}O_3$ VCLSUNMNDIKRMY-UHFFFAOYSA-N | $9.8\times10^3$ $6.0\times10^5$ $1.2\times10^3$ | | Wang et al. (2017) Wang et al. (2017) Wang et al. (2017) | Q Q Q | 80, 238 80, 239 80, 240 |
| MCM:C612OOH $C_6H_{12}O_4$ OANIXDZJWBEQCD-UHFFFAOYSA-N | $5.6\times10^6$ $9.3\times10^5$ $5.5\times10^3$ | | Wang et al. (2017) Wang et al. (2017) Wang et al. (2017) | Q Q Q | 80, 238 80, 239 80, 240 |





Table A3.6: Ketones (RCOR) (…continued)

| Substance Formula (Trivial Name) [CAS Registry Number] InChIKey | $H_s^{cp}$ (at $T^{\ominus}$) $\left[\dfrac{\text{mol}}{\text{m}^3\,\text{Pa}}\right]$ | $\dfrac{\text{d}\ln H_s^{cp}}{\text{d}(1/T)}$ [K] | Reference | Type | Note |
|---|---|---|---|---|---|
| MCM:C613OOH | $1.2\times10^9$ | | Wang et al. (2017) | Q | 80, 238 |
| $C_6H_{12}O_5$ | $2.0\times10^8$ | | Wang et al. (2017) | Q | 80, 239 |
| MBLADDFAKVGAFO-UHFFFAOYSA-N | $1.2\times10^4$ | | Wang et al. (2017) | Q | 80, 240 |
| MCM:C614CO | $2.5\times10^6$ | | Wang et al. (2017) | Q | 80, 238 |
| $C_6H_8O_4$ | $1.4\times10^6$ | | Wang et al. (2017) | Q | 80, 239 |
| DFWHGSVPVJPYJZ-UHFFFAOYSA-N | $8.9\times10^2$ | | Wang et al. (2017) | Q | 80, 240 |
| MCM:C614OH | $9.3\times10^7$ | | Wang et al. (2017) | Q | 80, 238 |
| $C_6H_{10}O_4$ | $9.1\times10^6$ | | Wang et al. (2017) | Q | 80, 239 |
| WLPHOQAEOJPJID-UHFFFAOYSA-N | $8.9\times10^3$ | | Wang et al. (2017) | Q | 80, 240 |
| MCM:C614OOH | $7.8\times10^9$ | | Wang et al. (2017) | Q | 80, 238 |
| $C_6H_{10}O_5$ | $8.3\times10^7$ | | Wang et al. (2017) | Q | 80, 239 |
| AZSLPKKGLPCKMI-UHFFFAOYSA-N | $4.2\times10^3$ | | Wang et al. (2017) | Q | 80, 240 |
| MCM:C619OH | $1.2\times10^4$ | | Wang et al. (2017) | Q | 80, 238 |
| $C_6H_8O_3$ | $2.5\times10^6$ | | Wang et al. (2017) | Q | 80, 239 |
| WEYIBABPZCYASX-UHFFFAOYSA-N | $4.9\times10^4$ | | Wang et al. (2017) | Q | 80, 240 |
| MCM:C61CO | $3.8\times10^3$ | | Wang et al. (2017) | Q | 80, 238 |
| $C_6H_{10}O_3$ | $2.2\times10^4$ | | Wang et al. (2017) | Q | 80, 239 |
| MGEAAKFVOQZVFH-UHFFFAOYSA-N | $1.0\times10^2$ | | Wang et al. (2017) | Q | 80, 240 |
| MCM:C61OH | $1.6\times10^5$ | | Wang et al. (2017) | Q | 80, 238 |
| $C_6H_{12}O_3$ | $3.6\times10^5$ | | Wang et al. (2017) | Q | 80, 239 |
| JOVKXRBMZIPRBZ-UHFFFAOYSA-N | $1.2\times10^3$ | | Wang et al. (2017) | Q | 80, 240 |
| MCM:C61OOH | $1.2\times10^7$ | | Wang et al. (2017) | Q | 80, 238 |
| $C_6H_{12}O_4$ | $1.7\times10^6$ | | Wang et al. (2017) | Q | 80, 239 |
| ZUQHNOSGBJLTRH-UHFFFAOYSA-N | $3.3\times10^3$ | | Wang et al. (2017) | Q | 80, 240 |
| MCM:C621OOH | $9.8\times10^{11}$ | | Wang et al. (2017) | Q | 80, 238 |
| $C_6H_{10}O_6$ | $2.7\times10^{10}$ | | Wang et al. (2017) | Q | 80, 239 |
| OMSKVAAPFLCUBC-UHFFFAOYSA-N | $3.5\times10^6$ | | Wang et al. (2017) | Q | 80, 240 |
| MCM:C624CO | $2.3\times10^2$ | | Wang et al. (2017) | Q | 80, 238 |
| $C_6H_{10}O_2$ | $6.3\times10^2$ | | Wang et al. (2017) | Q | 80, 239 |
| SAJQFIMRYKQCMA-UHFFFAOYSA-N | $5.6\times10^1$ | | Wang et al. (2017) | Q | 80, 240 |
| MCM:C625OH | $1.2\times10^7$ | | Wang et al. (2017) | Q | 80, 238 |
| $C_6H_{12}O_4$ | $3.9\times10^8$ | | Wang et al. (2017) | Q | 80, 239 |
| UHZLCNNVPSWLBR-UHFFFAOYSA-N | $2.1\times10^4$ | | Wang et al. (2017) | Q | 80, 240 |
| MCM:C625OOH | $2.0\times10^{10}$ | | Wang et al. (2017) | Q | 80, 238 |
| $C_6H_{12}O_5$ | $7.8\times10^8$ | | Wang et al. (2017) | Q | 80, 239 |
| RLEQEEKQTRHVBK-UHFFFAOYSA-N | $1.7\times10^6$ | | Wang et al. (2017) | Q | 80, 240 |
| MCM:C627OH | $3.7\times10^3$ | | Wang et al. (2017) | Q | 80, 238 |
| $C_6H_{10}O_3$ | $1.9\times10^5$ | | Wang et al. (2017) | Q | 80, 239 |
| YXADPHVQSSNJLB-UHFFFAOYSA-N | $5.0\times10^3$ | | Wang et al. (2017) | Q | 80, 240 |





Table A3.6: Ketones (RCOR) (...continued)

| Substance<br>Formula<br>(Trivial Name)<br>[CAS Registry Number]<br>InChIKey | $H_s^{cp}$<br>(at $T^\ominus$)<br>$\left[\dfrac{\mathrm{mol}}{\mathrm{m}^3\,\mathrm{Pa}}\right]$ | $\dfrac{\mathrm{d}\ln H_s^{cp}}{\mathrm{d}(1/T)}$<br><br>[K] | Reference | Type | Note |
|---|---|---|---|---|---|
| MCM:C63OOH<br>$C_6H_{10}O_3$<br>LTOZVYXSZHIEKZ-UHFFFAOYSA-N | $2.9\times10^4$<br>$1.7\times10^6$<br>$2.8\times10^4$ | | Wang et al. (2017)<br>Wang et al. (2017)<br>Wang et al. (2017) | Q<br>Q<br>Q | 80, 238<br>80, 239<br>80, 240 |
| MCM:C63OOOH<br>$C_6H_{10}O_4$<br>FPSNWVZQINIJKM-UHFFFAOYSA-N | $1.8\times10^7$<br>$2.2\times10^6$<br>$1.3\times10^5$ | | Wang et al. (2017)<br>Wang et al. (2017)<br>Wang et al. (2017) | Q<br>Q<br>Q | 80, 238<br>80, 239<br>80, 240 |
| MCM:C63OH<br>$C_6H_{12}O_3$<br>YEHSOHVSOJHCML-UHFFFAOYSA-N | $1.4\times10^5$<br>$1.2\times10^5$<br>$2.0\times10^3$ | | Wang et al. (2017)<br>Wang et al. (2017)<br>Wang et al. (2017) | Q<br>Q<br>Q | 80, 238<br>80, 239<br>80, 240 |
| MCM:C63OOH<br>$C_6H_{12}O_4$<br>RYUSCYCDUHZJRL-UHFFFAOYSA-N | $1.0\times10^7$<br>$1.1\times10^6$<br>$1.2\times10^4$ | | Wang et al. (2017)<br>Wang et al. (2017)<br>Wang et al. (2017) | Q<br>Q<br>Q | 80, 238<br>80, 239<br>80, 240 |
| MCM:C64OH<br>$C_6H_{12}O_3$<br>DWHFPVYPGLVOER-UHFFFAOYSA-N | $8.5\times10^4$<br>$2.2\times10^5$<br>$1.1\times10^3$ | | Wang et al. (2017)<br>Wang et al. (2017)<br>Wang et al. (2017) | Q<br>Q<br>Q | 80, 238<br>80, 239<br>80, 240 |
| MCM:C64OOH<br>$C_6H_{12}O_4$<br>WKWKMZOJIHCBFC-UHFFFAOYSA-N | $7.3\times10^6$<br>$1.1\times10^6$<br>$1.8\times10^3$ | | Wang et al. (2017)<br>Wang et al. (2017)<br>Wang et al. (2017) | Q<br>Q<br>Q | 80, 238<br>80, 239<br>80, 240 |
| MCM:C66CO<br>$C_6H_{10}O_3$<br>PQXOVUFKUOIXAY-UHFFFAOYSA-N | $7.4\times10^4$<br>$2.9\times10^4$<br>$4.2\times10^1$ | | Wang et al. (2017)<br>Wang et al. (2017)<br>Wang et al. (2017) | Q<br>Q<br>Q | 80, 238<br>80, 239<br>80, 240 |
| MCM:C66OH<br>$C_6H_{12}O_3$<br>YKCRSYHGTATDJD-UHFFFAOYSA-N | $1.7\times10^4$<br>$9.3\times10^5$<br>$6.2\times10^3$ | | Wang et al. (2017)<br>Wang et al. (2017)<br>Wang et al. (2017) | Q<br>Q<br>Q | 80, 238<br>80, 239<br>80, 240 |
| MCM:C66OOH<br>$C_6H_{12}O_4$<br>ZIMKSIUZUDHWGV-UHFFFAOYSA-N | $1.0\times10^7$<br>$7.4\times10^5$<br>$9.3\times10^4$ | | Wang et al. (2017)<br>Wang et al. (2017)<br>Wang et al. (2017) | Q<br>Q<br>Q | 80, 238<br>80, 239<br>80, 240 |
| MCM:C67CO3H<br>$C_6H_{10}O_5$<br>BGZGHPIMJSIBNB-UHFFFAOYSA-N | $3.9\times10^6$<br>$6.6\times10^5$<br>$3.4\times10^2$ | | Wang et al. (2017)<br>Wang et al. (2017)<br>Wang et al. (2017) | Q<br>Q<br>Q | 80, 238<br>80, 239<br>80, 240 |
| MCM:C69OH<br>$C_6H_{12}O_3$<br>UAFCVHAXODPLTF-UHFFFAOYSA-N | $1.7\times10^4$<br>$8.9\times10^5$<br>$2.0\times10^3$ | | Wang et al. (2017)<br>Wang et al. (2017)<br>Wang et al. (2017) | Q<br>Q<br>Q | 80, 238<br>80, 239<br>80, 240 |
| MCM:C69OOH<br>$C_6H_{12}O_4$<br>IBTXITPBBCYNLS-UHFFFAOYSA-N | $1.0\times10^7$<br>$1.3\times10^6$<br>$6.8\times10^3$ | | Wang et al. (2017)<br>Wang et al. (2017)<br>Wang et al. (2017) | Q<br>Q<br>Q | 80, 238<br>80, 239<br>80, 240 |
| MCM:C6CO23HO5<br>$C_6H_{10}O_3$<br>ZFXUWHPHUNDYBV-UHFFFAOYSA-N | $7.4\times10^4$<br>$2.6\times10^4$<br>$5.6\times10^1$ | | Wang et al. (2017)<br>Wang et al. (2017)<br>Wang et al. (2017) | Q<br>Q<br>Q | 80, 238<br>80, 239<br>80, 240 |





Table A3.6: Ketones (RCOR) (...continued)

| Substance Formula (Trivial Name) [CAS Registry Number] InChIKey | $H_s^{cp}$ (at $T^{\ominus}$) $\left[\dfrac{\text{mol}}{\text{m}^3\,\text{Pa}}\right]$ | $\dfrac{\text{d}\ln H_s^{cp}}{\text{d}(1/T)}$ [K] | Reference | Type | Note |
|---|---|---|---|---|---|
| MCM:C6CO243OH | $8.5\times10^2$ | | Wang et al. (2017) | Q | 80, 238 |
| $C_6H_{10}O_3$ | $6.6\times10^3$ | | Wang et al. (2017) | Q | 80, 239 |
| YZRCADLLOAXRLD-UHFFFAOYSA-N | 5.0 | | Wang et al. (2017) | Q | 80, 240 |
| MCM:C6CO34HO1 | $7.3\times10^4$ | | Wang et al. (2017) | Q | 80, 238 |
| $C_6H_{10}O_3$ | $2.0\times10^4$ | | Wang et al. (2017) | Q | 80, 239 |
| LBGWMVXVPMPEID-UHFFFAOYSA-N | $9.3\times10^1$ | | Wang et al. (2017) | Q | 80, 240 |
| MCM:C6CO3HO14 | $1.6\times10^4$ | | Wang et al. (2017) | Q | 80, 238 |
| $C_6H_{12}O_3$ | $9.1\times10^5$ | | Wang et al. (2017) | Q | 80, 239 |
| AEYYIRCLAOJEMX-UHFFFAOYSA-N | $9.1\times10^2$ | | Wang et al. (2017) | Q | 80, 240 |
| MCM:C6CO3HO25 | $1.7\times10^4$ | | Wang et al. (2017) | Q | 80, 238 |
| $C_6H_{12}O_3$ | $1.1\times10^6$ | | Wang et al. (2017) | Q | 80, 239 |
| AQJAHIKLRWQRAX-UHFFFAOYSA-N | $1.8\times10^3$ | | Wang et al. (2017) | Q | 80, 240 |
| MCM:C6CO3HO4 | 5.6 | | Wang et al. (2017) | Q | 80, 238 |
| $C_6H_{12}O_2$ | $4.6\times10^1$ | | Wang et al. (2017) | Q | 80, 239 |
| SKCYVGUCBRYGTE-UHFFFAOYSA-N | $6.2\times10^2$ | | Wang et al. (2017) | Q | 80, 240 |
| MCM:C6CO3OHOOH | $1.0\times10^7$ | | Wang et al. (2017) | Q | 80, 238 |
| $C_6H_{12}O_4$ | $6.3\times10^5$ | | Wang et al. (2017) | Q | 80, 239 |
| VJHRZIUXRMRGDC-UHFFFAOYSA-N | $6.5\times10^4$ | | Wang et al. (2017) | Q | 80, 240 |
| MCM:C6HOCOOOH | $9.1\times10^6$ | | Wang et al. (2017) | Q | 80, 238 |
| $C_6H_{12}O_4$ | $7.4\times10^5$ | | Wang et al. (2017) | Q | 80, 239 |
| DEHZHOWOLMWYLS-UHFFFAOYSA-N | $8.0\times10^4$ | | Wang et al. (2017) | Q | 80, 240 |
| MCM:CO24M3C5OH | $3.8\times10^3$ | | Wang et al. (2017) | Q | 80, 238 |
| $C_6H_{10}O_3$ | $1.1\times10^4$ | | Wang et al. (2017) | Q | 80, 239 |
| JUTFYFRULBWUNH-UHFFFAOYSA-N | $2.1\times10^2$ | | Wang et al. (2017) | Q | 80, 240 |
| MCM:CO25C6OH | $3.8\times10^3$ | | Wang et al. (2017) | Q | 80, 238 |
| $C_6H_{10}O_3$ | $1.6\times10^5$ | | Wang et al. (2017) | Q | 80, 239 |
| MXGSILCFPNVDMX-UHFFFAOYSA-N | $2.0\times10^3$ | | Wang et al. (2017) | Q | 80, 240 |
| MCM:CO2HO3C6 | 5.6 | | Wang et al. (2017) | Q | 80, 238 |
| $C_6H_{12}O_2$ | $5.8\times10^1$ | | Wang et al. (2017) | Q | 80, 239 |
| UHSBCAJZDUQTHH-UHFFFAOYSA-N | 4.1 | | Wang et al. (2017) | Q | 80, 240 |
| MCM:CO2HO4C6 | $1.1\times10^2$ | | Wang et al. (2017) | Q | 80, 238 |
| $C_6H_{12}O_2$ | $7.8\times10^2$ | | Wang et al. (2017) | Q | 80, 239 |
| ODWYTDVNWFBCLV-UHFFFAOYSA-N | $3.9\times10^1$ | | Wang et al. (2017) | Q | 80, 240 |
| MCM:CO2HOC6OOH | $9.1\times10^6$ | | Wang et al. (2017) | Q | 80, 238 |
| $C_6H_{12}O_4$ | $1.0\times10^6$ | | Wang et al. (2017) | Q | 80, 239 |
| DONIGPAOCGSEHJ-UHFFFAOYSA-N | $1.3\times10^4$ | | Wang et al. (2017) | Q | 80, 240 |
| MCM:CO2M3C5OH | $1.1\times10^2$ | | Wang et al. (2017) | Q | 80, 238 |
| $C_6H_{12}O_2$ | $1.7\times10^3$ | | Wang et al. (2017) | Q | 80, 239 |
| YJCMJVMCCJIPRV-UHFFFAOYSA-N | $4.6\times10^2$ | | Wang et al. (2017) | Q | 80, 240 |



Table A3.6: Ketones (RCOR) (...continued)

| Substance<br>Formula<br>(Trivial Name)<br>[CAS Registry Number]<br>InChIKey | $H_s^{cp}$<br>(at $T^{\ominus}$)<br>$\left[\dfrac{\text{mol}}{\text{m}^3\,\text{Pa}}\right]$ | $\dfrac{\text{d}\ln H_s^{cp}}{\text{d}(1/T)}$<br><br>[K] | Reference | Type | Note |
|---|---|---|---|---|---|
| MCM:CO2MC5OH | $1.1\times10^2$ | | Wang et al. (2017) | Q | 80, 238 |
| $C_6H_{12}O_2$ | $1.9\times10^3$ | | Wang et al. (2017) | Q | 80, 239 |
| NZBRXFKHZBOFBW-UHFFFAOYSA-N | $5.1\times10^2$ | | Wang et al. (2017) | Q | 80, 240 |
| MCM:CY6DIONOH | $2.3\times10^5$ | | Wang et al. (2017) | Q | 80, 238 |
| $C_6H_8O_3$ | $6.5\times10^7$ | | Wang et al. (2017) | Q | 80, 239 |
| ACRUCDPQQCIOID-UHFFFAOYSA-N | $2.9\times10^7$ | | Wang et al. (2017) | Q | 80, 240 |
| MCM:CYHXOLACO | $1.8\times10^1$ | | Wang et al. (2017) | Q | 80, 238 |
| $C_6H_{10}O_2$ | $1.4\times10^3$ | | Wang et al. (2017) | Q | 80, 239 |
| ODZTXUXIYGJLMC-UHFFFAOYSA-N | $6.8\times10^1$ | | Wang et al. (2017) | Q | 80, 240 |
| MCM:CYHXONAOH | $3.5\times10^2$ | | Wang et al. (2017) | Q | 80, 238 |
| $C_6H_{10}O_2$ | $3.5\times10^4$ | | Wang et al. (2017) | Q | 80, 239 |
| TWEVQGUWCLBRMJ-UHFFFAOYSA-N | $8.7\times10^4$ | | Wang et al. (2017) | Q | 80, 240 |
| MCM:DMK2OH | $2.8\times10^5$ | | Wang et al. (2017) | Q | 80, 238 |
| $C_6H_{10}O_4$ | $4.2\times10^6$ | | Wang et al. (2017) | Q | 80, 239 |
| RQDWELNLPMBYMA-UHFFFAOYSA-N | $2.6\times10^2$ | | Wang et al. (2017) | Q | 80, 240 |
| MCM:DMKCOOH | $6.5\times10^5$ | | Wang et al. (2017) | Q | 80, 238 |
| $C_6H_8O_4$ | $3.2\times10^5$ | | Wang et al. (2017) | Q | 80, 239 |
| OKDWWGVTKQPUTM-UHFFFAOYSA-N | $1.0\times10^1$ | | Wang et al. (2017) | Q | 80, 240 |
| MCM:DMKOHOOH | $4.1\times10^8$ | | Wang et al. (2017) | Q | 80, 238 |
| $C_6H_{10}O_5$ | $7.3\times10^6$ | | Wang et al. (2017) | Q | 80, 239 |
| WAHPQVAXIMVNLL-UHFFFAOYSA-N | $1.0\times10^4$ | | Wang et al. (2017) | Q | 80, 240 |
| MCM:EIPKAOH | $3.9$ | | Wang et al. (2017) | Q | 80, 238 |
| $C_6H_{12}O_2$ | $3.3\times10^1$ | | Wang et al. (2017) | Q | 80, 239 |
| SYAVYWAMVZKGTF-UHFFFAOYSA-N | $4.5$ | | Wang et al. (2017) | Q | 80, 240 |
| MCM:EIPKBOH | $1.1\times10^2$ | | Wang et al. (2017) | Q | 80, 238 |
| $C_6H_{12}O_2$ | $3.1\times10^2$ | | Wang et al. (2017) | Q | 80, 239 |
| JYEMLYGXHDIVRA-UHFFFAOYSA-N | $4.9\times10^1$ | | Wang et al. (2017) | Q | 80, 240 |
| MCM:H13M3CO4C5 | $9.8\times10^3$ | | Wang et al. (2017) | Q | 80, 238 |
| $C_6H_{12}O_3$ | $1.7\times10^5$ | | Wang et al. (2017) | Q | 80, 239 |
| NBNMOYHUQSLZCQ-UHFFFAOYSA-N | $3.5\times10^3$ | | Wang et al. (2017) | Q | 80, 240 |
| MCM:H3C25C6OH | $5.6\times10^5$ | | Wang et al. (2017) | Q | 80, 238 |
| $C_6H_{10}O_4$ | $4.8\times10^7$ | | Wang et al. (2017) | Q | 80, 239 |
| XLUGOLRJPRATMY-UHFFFAOYSA-N | $3.5\times10^4$ | | Wang et al. (2017) | Q | 80, 240 |
| MCM:H3C25C6OOH | $2.9\times10^8$ | | Wang et al. (2017) | Q | 80, 238 |
| $C_6H_{10}O_5$ | $5.0\times10^7$ | | Wang et al. (2017) | Q | 80, 239 |
| XIJKANBCMKWKRM-UHFFFAOYSA-N | $1.8\times10^4$ | | Wang et al. (2017) | Q | 80, 240 |
| MCM:HEX3ONAOH | $1.1\times10^2$ | | Wang et al. (2017) | Q | 80, 238 |
| $C_6H_{12}O_2$ | $6.5\times10^2$ | | Wang et al. (2017) | Q | 80, 239 |
| XHYXKWWFNRBCGE-UHFFFAOYSA-N | $4.2\times10^1$ | | Wang et al. (2017) | Q | 80, 240 |





Table A3.6: Ketones (RCOR) (...continued)

| Substance Formula (Trivial Name) [CAS Registry Number] InChIKey | $H_s^{cp}$ (at $T^{\ominus}$) $\left[\dfrac{\text{mol}}{\text{m}^3\,\text{Pa}}\right]$ | $\dfrac{\text{d}\ln H_s^{cp}}{\text{d}(1/T)}$ [K] | Reference | Type | Note |
|---|---|---|---|---|---|
| MCM:HEX3ONCOH $C_6H_{12}O_2$ ZWBUSAWJHMPOEJ-UHFFFAOYSA-N | 5.6 $4.2\times10^1$ 3.5 | | Wang et al. (2017) Wang et al. (2017) Wang et al. (2017) | Q Q Q | 80, 238 80, 239 80, 240 |
| MCM:HEX3ONDOH $C_6H_{12}O_2$ ITHSWIXXHGKFJW-UHFFFAOYSA-N | $9.8\times10^1$ $2.6\times10^2$ $5.0\times10^1$ | | Wang et al. (2017) Wang et al. (2017) Wang et al. (2017) | Q Q Q | 80, 238 80, 239 80, 240 |
| MCM:HO1CO24C6 $C_6H_{10}O_3$ NUDMYLMVZRZHHU-UHFFFAOYSA-N | $3.7\times10^3$ $1.4\times10^4$ $1.5\times10^2$ | | Wang et al. (2017) Wang et al. (2017) Wang et al. (2017) | Q Q Q | 80, 238 80, 239 80, 240 |
| MCM:HO1CO4C6 $C_6H_{12}O_2$ APQMHEQLBDXGMP-UHFFFAOYSA-N | $9.8\times10^1$ $1.4\times10^3$ $4.8\times10^2$ | | Wang et al. (2017) Wang et al. (2017) Wang et al. (2017) | Q Q Q | 80, 238 80, 239 80, 240 |
| MCM:HO2CO5C6 $C_6H_{12}O_2$ ZSDLLTJVENEIDW-UHFFFAOYSA-N | $2.8\times10^3$ $1.1\times10^2$ $2.3\times10^3$ $6.9\times10^2$ | 10000 | Wieser et al. (2023) Wang et al. (2017) Wang et al. (2017) Wang et al. (2017) | Q Q Q Q | 437 80, 238 80, 239 80, 240 |
| MCM:M2BKAOH $C_6H_{12}O_2$ ZXZUCILLTLHIBZ-UHFFFAOYSA-N | $1.3\times10^2$ $6.9\times10^2$ $4.6\times10^1$ | | Wang et al. (2017) Wang et al. (2017) Wang et al. (2017) | Q Q Q | 80, 238 80, 239 80, 240 |
| MCM:M2BKBOH $C_6H_{12}O_2$ KHCUSEDRQWYNDS-UHFFFAOYSA-N | 3.9 $4.4\times10^1$ 3.6 | | Wang et al. (2017) Wang et al. (2017) Wang et al. (2017) | Q Q Q | 80, 238 80, 239 80, 240 |
| MCM:MBKCOOHOOH $C_6H_{10}O_5$ VGLNQNYMXFMHPV-UHFFFAOYSA-N | $2.5\times10^8$ $2.1\times10^6$ $4.6\times10^2$ | | Wang et al. (2017) Wang et al. (2017) Wang et al. (2017) | Q Q Q | 80, 238 80, 239 80, 240 |
| MCM:MIBKAOH3CO $C_6H_{10}O_3$ NARMPYMUEZMSEV-UHFFFAOYSA-N | $2.6\times10^3$ $4.6\times10^2$ 2.4 | | Wang et al. (2017) Wang et al. (2017) Wang et al. (2017) | Q Q Q | 80, 238 80, 239 80, 240 |
| MCM:MIBKBOH $C_6H_{12}O_2$ IGPIDYBTABPKQT-UHFFFAOYSA-N | 6.5 $6.3\times10^1$ 2.6 | | Wang et al. (2017) Wang et al. (2017) Wang et al. (2017) | Q Q Q | 80, 238 80, 239 80, 240 |
| MCM:MIBKHO14 $C_6H_{12}O_3$ UZSRJTNVDUYOLE-UHFFFAOYSA-N | $9.8\times10^3$ $8.1\times10^5$ $2.3\times10^3$ | | Wang et al. (2017) Wang et al. (2017) Wang et al. (2017) | Q Q Q | 80, 238 80, 239 80, 240 |
| MCM:MIBKHO4OOH $C_6H_{12}O_4$ FGHCAVJLMYVGOV-UHFFFAOYSA-N | $5.6\times10^6$ $1.1\times10^6$ $1.7\times10^4$ | | Wang et al. (2017) Wang et al. (2017) Wang et al. (2017) | Q Q Q | 80, 238 80, 239 80, 240 |
| MCM:MIBKOH34 $C_6H_{12}O_3$ OSPIOFUQWMUWOI-UHFFFAOYSA-N | $5.0\times10^3$ $4.1\times10^4$ $3.2\times10^2$ | | Wang et al. (2017) Wang et al. (2017) Wang et al. (2017) | Q Q Q | 80, 238 80, 239 80, 240 |





Table A3.6: Ketones (RCOR) (. . . continued)

| Substance Formula (Trivial Name) [CAS Registry Number] InChIKey | $H_s^{cp}$ (at $T^\ominus$) $\left[\dfrac{\text{mol}}{\text{m}^3\,\text{Pa}}\right]$ | $\dfrac{\text{d}\ln H_s^{cp}}{\text{d}(1/T)}$ [K] | Reference | Type | Note |
|---|---|---|---|---|---|
| MCM:MIBKOHAOOH | $8.1\times10^6$ | | Wang et al. (2017) | Q | 80, 238 |
| $C_6H_{12}O_4$ | $1.8\times10^5$ | | Wang et al. (2017) | Q | 80, 239 |
| WIXFQYMFSUQGGT-UHFFFAOYSA-N | $4.8\times10^2$ | | Wang et al. (2017) | Q | 80, 240 |
| MCM:MIBKOHBOOH | $7.3\times10^6$ | | Wang et al. (2017) | Q | 80, 238 |
| $C_6H_{12}O_4$ | $2.9\times10^5$ | | Wang et al. (2017) | Q | 80, 239 |
| NCKNZQNPSVRYAP-UHFFFAOYSA-N | $3.2\times10^3$ | | Wang et al. (2017) | Q | 80, 240 |
| MCM:MTBKOH | $7.6\times10^1$ | | Wang et al. (2017) | Q | 80, 238 |
| $C_6H_{12}O_2$ | $3.8\times10^2$ | | Wang et al. (2017) | Q | 80, 239 |
| DYAWMXSWDGPGOI-UHFFFAOYSA-N | $3.2\times10^1$ | | Wang et al. (2017) | Q | 80, 240 |
| MCM:PBZQCO | $8.3\times10^6$ | | Wang et al. (2017) | Q | 80, 238 |
| $C_6H_4O_4$ | $3.6\times10^9$ | | Wang et al. (2017) | Q | 80, 239 |
| LTWQMQDFZHUKCN-UHFFFAOYSA-N | $3.4\times10^5$ | | Wang et al. (2017) | Q | 80, 240 |
| MCM:PBZQOH | $3.6\times10^6$ | | Wang et al. (2017) | Q | 80, 238 |
| $C_6H_6O_4$ | $5.0\times10^8$ | | Wang et al. (2017) | Q | 80, 239 |
| KGAUAKJVBKFEJK-UHFFFAOYSA-N | $1.5\times10^5$ | | Wang et al. (2017) | Q | 80, 240 |
| MCM:PBZQOOH | $5.3\times10^9$ | | Wang et al. (2017) | Q | 80, 238 |
| $C_6H_6O_5$ | $1.9\times10^9$ | | Wang et al. (2017) | Q | 80, 239 |
| GWWKDNIKIVTFLV-UHFFFAOYSA-N | $1.4\times10^6$ | | Wang et al. (2017) | Q | 80, 240 |
| MCM:C61CO3H | $7.8\times10^8$ | | Wang et al. (2017) | Q | 80, 238 |
| $C_7H_{10}O_6$ | $4.6\times10^5$ | | Wang et al. (2017) | Q | 80, 239 |
| LNQZNTUCDYUQNY-UHFFFAOYSA-N | $4.1\times10^2$ | | Wang et al. (2017) | Q | 80, 240 |
| MCM:C62CO3H | $8.9\times10^8$ | | Wang et al. (2017) | Q | 80, 238 |
| $C_7H_{10}O_6$ | $5.1\times10^5$ | | Wang et al. (2017) | Q | 80, 239 |
| OTMFEIRKQNFPTO-UHFFFAOYSA-N | $1.7\times10^2$ | | Wang et al. (2017) | Q | 80, 240 |
| MCM:C711OOH | $5.0\times10^6$ | | Wang et al. (2017) | Q | 80, 238 |
| $C_7H_{14}O_4$ | $7.4\times10^5$ | | Wang et al. (2017) | Q | 80, 239 |
| CHMPWQJFRSDBMP-UHFFFAOYSA-N | $5.9\times10^3$ | | Wang et al. (2017) | Q | 80, 240 |
| MCM:C712OH | $8.7\times10^4$ | | Wang et al. (2017) | Q | 80, 238 |
| $C_7H_{14}O_3$ | $3.6\times10^5$ | | Wang et al. (2017) | Q | 80, 239 |
| XKQDJNZCXUTPAB-UHFFFAOYSA-N | $5.5\times10^2$ | | Wang et al. (2017) | Q | 80, 240 |
| MCM:C712OOH | $6.5\times10^6$ | | Wang et al. (2017) | Q | 80, 238 |
| $C_7H_{14}O_4$ | $7.6\times10^5$ | | Wang et al. (2017) | Q | 80, 239 |
| ISDXEEIPAJYQKS-UHFFFAOYSA-N | $8.0\times10^2$ | | Wang et al. (2017) | Q | 80, 240 |
| MCM:C713OH | $2.1\times10^3$ | | Wang et al. (2017) | Q | 80, 238 |
| $C_7H_{12}O_3$ | $4.0\times10^4$ | | Wang et al. (2017) | Q | 80, 239 |
| KKMPMNHIJNYFMY-UHFFFAOYSA-N | $2.7\times10^2$ | | Wang et al. (2017) | Q | 80, 240 |
| MCM:C714OH | $3.6\times10^3$ | | Wang et al. (2017) | Q | 80, 238 |
| $C_7H_{12}O_3$ | $6.8\times10^4$ | | Wang et al. (2017) | Q | 80, 239 |
| BDJIDAHXOAZMLT-UHFFFAOYSA-N | $4.2\times10^2$ | | Wang et al. (2017) | Q | 80, 240 |



Table A3.6: Ketones (RCOR) (... continued)

| Substance<br>Formula<br>(Trivial Name)<br>[CAS Registry Number]<br>InChIKey | $H_s^{cp}$<br>(at $T^{\ominus}$)<br>$\left[\dfrac{\mathrm{mol}}{\mathrm{m^3\,Pa}}\right]$ | $\dfrac{\mathrm{d}\ln H_s^{cp}}{\mathrm{d}(1/T)}$<br><br>[K] | Reference | Type | Note |
|---|---|---|---|---|---|
| MCM:C719OH | $5.4\times10^8$ | | Wang et al. (2017) | Q | 80, 238 |
| $C_7H_{12}O_4$ | $7.1\times10^{10}$ | | Wang et al. (2017) | Q | 80, 239 |
| SNLGDBZBHBGDGF-UHFFFAOYSA-N | $2.3\times10^8$ | | Wang et al. (2017) | Q | 80, 240 |
| MCM:C719OOH | $7.1\times10^{10}$ | | Wang et al. (2017) | Q | 80, 238 |
| $C_7H_{12}O_5$ | $5.8\times10^{11}$ | | Wang et al. (2017) | Q | 80, 239 |
| RIQZESWBZQGTOF-UHFFFAOYSA-N | $2.0\times10^9$ | | Wang et al. (2017) | Q | 80, 240 |
| MCM:C71OOH | $8.5\times10^6$ | | Wang et al. (2017) | Q | 80, 238 |
| $C_7H_{14}O_4$ | $4.1\times10^5$ | | Wang et al. (2017) | Q | 80, 239 |
| NBQALOYIKVIATH-UHFFFAOYSA-N | $3.3\times10^4$ | | Wang et al. (2017) | Q | 80, 240 |
| MCM:C726CO3OH | $1.4\times10^4$ | | Wang et al. (2017) | Q | 80, 238 |
| $C_7H_{10}O_3$ | $9.1\times10^4$ | | Wang et al. (2017) | Q | 80, 239 |
| KXKCGTAVIRFZLI-UHFFFAOYSA-N | $6.0\times10^3$ | | Wang et al. (2017) | Q | 80, 240 |
| MCM:C72OH | $1.3\times10^5$ | | Wang et al. (2017) | Q | 80, 238 |
| $C_7H_{14}O_3$ | $1.9\times10^5$ | | Wang et al. (2017) | Q | 80, 239 |
| NSVUCXUEJIZYCX-UHFFFAOYSA-N | $8.7\times10^2$ | | Wang et al. (2017) | Q | 80, 240 |
| MCM:C72OOH | $9.3\times10^6$ | | Wang et al. (2017) | Q | 80, 238 |
| $C_7H_{14}O_4$ | $9.3\times10^5$ | | Wang et al. (2017) | Q | 80, 239 |
| HOCRXLJRXJDOAI-UHFFFAOYSA-N | $1.8\times10^3$ | | Wang et al. (2017) | Q | 80, 240 |
| MCM:C76OOH | $8.5\times10^6$ | | Wang et al. (2017) | Q | 80, 238 |
| $C_7H_{14}O_4$ | $8.0\times10^5$ | | Wang et al. (2017) | Q | 80, 239 |
| RNKPJWWQPAQZAU-UHFFFAOYSA-N | $1.7\times10^4$ | | Wang et al. (2017) | Q | 80, 240 |
| MCM:C77CO | $2.1\times10^3$ | | Wang et al. (2017) | Q | 80, 238 |
| $C_7H_{12}O_3$ | $8.1\times10^3$ | | Wang et al. (2017) | Q | 80, 239 |
| LRWMVVHMPWURQP-UHFFFAOYSA-N | $7.3\times10^1$ | | Wang et al. (2017) | Q | 80, 240 |
| MCM:C77OH | $8.7\times10^4$ | | Wang et al. (2017) | Q | 80, 238 |
| $C_7H_{14}O_3$ | $3.2\times10^5$ | | Wang et al. (2017) | Q | 80, 239 |
| BWFQHNYIFKCYIN-UHFFFAOYSA-N | $8.7\times10^2$ | | Wang et al. (2017) | Q | 80, 240 |
| MCM:C77OOH | $6.5\times10^6$ | | Wang et al. (2017) | Q | 80, 238 |
| $C_7H_{14}O_4$ | $6.5\times10^5$ | | Wang et al. (2017) | Q | 80, 239 |
| VLZZBCPZVYCEGU-UHFFFAOYSA-N | $4.5\times10^3$ | | Wang et al. (2017) | Q | 80, 240 |
| MCM:C78CO | $4.2\times10^4$ | | Wang et al. (2017) | Q | 80, 238 |
| $C_7H_{12}O_3$ | $1.8\times10^4$ | | Wang et al. (2017) | Q | 80, 239 |
| XZDSUXRHWGXGLP-UHFFFAOYSA-N | $2.8\times10^1$ | | Wang et al. (2017) | Q | 80, 240 |
| MCM:C78OH | $9.1\times10^3$ | | Wang et al. (2017) | Q | 80, 238 |
| $C_7H_{14}O_3$ | $3.0\times10^5$ | | Wang et al. (2017) | Q | 80, 239 |
| RHPFPSWJNNLJIH-UHFFFAOYSA-N | $1.6\times10^3$ | | Wang et al. (2017) | Q | 80, 240 |
| MCM:C78OOH | $5.6\times10^6$ | | Wang et al. (2017) | Q | 80, 238 |
| $C_7H_{14}O_4$ | $3.7\times10^5$ | | Wang et al. (2017) | Q | 80, 239 |
| CHSMMMZNDOATLZ-UHFFFAOYSA-N | $1.4\times10^4$ | | Wang et al. (2017) | Q | 80, 240 |





Table A3.6: Ketones (RCOR) (...continued)

| Substance Formula (Trivial Name) [CAS Registry Number] InChIKey | $H_s^{cp}$ (at $T^{\ominus}$) $\left[\dfrac{\mathrm{mol}}{\mathrm{m^3\,Pa}}\right]$ | $\dfrac{\mathrm{d}\ln H_s^{cp}}{\mathrm{d}(1/T)}$ [K] | Reference | Type | Note |
|---|---|---|---|---|---|
| MCM:C79OOH | $2.2\times10^8$ | | Wang et al. (2017) | Q | 80, 238 |
| $C_7H_{12}O_5$ | $1.0\times10^7$ | | Wang et al. (2017) | Q | 80, 239 |
| PAASZTKJDWJXEO-UHFFFAOYSA-N | $1.7\times10^3$ | | Wang et al. (2017) | Q | 80, 240 |
| MCM:C7BDCOH | $1.6\times10^5$ | | Wang et al. (2017) | Q | 80, 238 |
| $C_7H_{12}O_4$ | $3.6\times10^6$ | | Wang et al. (2017) | Q | 80, 239 |
| IQLAANIUQOXZGT-UHFFFAOYSA-N | $7.1\times10^1$ | | Wang et al. (2017) | Q | 80, 240 |
| MCM:C7BDCOOH | $2.3\times10^8$ | | Wang et al. (2017) | Q | 80, 238 |
| $C_7H_{12}O_5$ | $2.5\times10^6$ | | Wang et al. (2017) | Q | 80, 239 |
| YSGKLAZJXASOJQ-UHFFFAOYSA-N | $2.9\times10^1$ | | Wang et al. (2017) | Q | 80, 240 |
| MCM:C7EDCCO | $5.3\times10^5$ | | Wang et al. (2017) | Q | 80, 238 |
| $C_7H_{10}O_4$ | $2.2\times10^5$ | | Wang et al. (2017) | Q | 80, 239 |
| ZZJOBVIMQADRSG-UHFFFAOYSA-N | 8.7 | | Wang et al. (2017) | Q | 80, 240 |
| MCM:C7EDCOH | $2.3\times10^5$ | | Wang et al. (2017) | Q | 80, 238 |
| $C_7H_{12}O_4$ | $2.7\times10^6$ | | Wang et al. (2017) | Q | 80, 239 |
| ILHMXLYATWSMQV-UHFFFAOYSA-N | $1.7\times10^2$ | | Wang et al. (2017) | Q | 80, 240 |
| MCM:C7EDCOOH | $3.7\times10^8$ | | Wang et al. (2017) | Q | 80, 238 |
| $C_7H_{12}O_5$ | $4.6\times10^6$ | | Wang et al. (2017) | Q | 80, 239 |
| GFYCWCYSZRNKIC-UHFFFAOYSA-N | $4.3\times10^3$ | | Wang et al. (2017) | Q | 80, 240 |
| MCM:CO25C73OH | $3.5\times10^3$ | | Wang et al. (2017) | Q | 80, 238 |
| $C_7H_{12}O_3$ | $8.3\times10^4$ | | Wang et al. (2017) | Q | 80, 239 |
| SUAOXERUOLNODJ-UHFFFAOYSA-N | $1.0\times10^3$ | | Wang et al. (2017) | Q | 80, 240 |
| MCM:CO25C74OH | $3.5\times10^3$ | | Wang et al. (2017) | Q | 80, 238 |
| $C_7H_{12}O_3$ | $8.5\times10^4$ | | Wang et al. (2017) | Q | 80, 239 |
| WVDVIEDFZYMNAI-UHFFFAOYSA-N | $1.2\times10^3$ | | Wang et al. (2017) | Q | 80, 240 |
| MCM:H2M2CO5C6 | $6.0\times10^1$ | | Wang et al. (2017) | Q | 80, 238 |
| $C_7H_{14}O_2$ | $1.5\times10^3$ | | Wang et al. (2017) | Q | 80, 239 |
| GNNBSAPGFGNCCT-UHFFFAOYSA-N | $4.2\times10^2$ | | Wang et al. (2017) | Q | 80, 240 |
| MCM:H2M3CO5C6 | $1.0\times10^2$ | | Wang et al. (2017) | Q | 80, 238 |
| $C_7H_{14}O_2$ | $1.7\times10^3$ | | Wang et al. (2017) | Q | 80, 239 |
| JXXNGUYHZPNXHL-UHFFFAOYSA-N | $2.4\times10^2$ | | Wang et al. (2017) | Q | 80, 240 |
| MCM:H2M4CO5C6 | $1.0\times10^2$ | | Wang et al. (2017) | Q | 80, 238 |
| $C_7H_{14}O_2$ | $1.4\times10^3$ | | Wang et al. (2017) | Q | 80, 239 |
| ZIVAMUMQDXZFCR-UHFFFAOYSA-N | $2.3\times10^2$ | | Wang et al. (2017) | Q | 80, 240 |
| MCM:H3C25CCO3H | $3.3\times10^9$ | | Wang et al. (2017) | Q | 80, 238 |
| $C_7H_{10}O_6$ | $9.8\times10^8$ | | Wang et al. (2017) | Q | 80, 239 |
| ARDZNQBCJILRHG-UHFFFAOYSA-N | $6.0\times10^4$ | | Wang et al. (2017) | Q | 80, 240 |
| MCM:HCC7CO | $9.1\times10^2$ | | Wang et al. (2017) | Q | 80, 238 |
| $C_7H_{10}O_2$ | $4.6\times10^4$ | | Wang et al. (2017) | Q | 80, 239 |
| WVMVYBISUAVCBD-UHFFFAOYSA-N | $1.9\times10^4$ | | Wang et al. (2017) | Q | 80, 240 |





Table A3.6: Ketones (RCOR) (...continued)

| Substance Formula (Trivial Name) [CAS Registry Number] InChIKey | $H_s^{cp}$ (at $T^{\ominus}$) $\left[\dfrac{\mathrm{mol}}{\mathrm{m^3\,Pa}}\right]$ | $\dfrac{\mathrm{d}\ln H_s^{cp}}{\mathrm{d}(1/T)}$ [K] | Reference | Type | Note |
|---|---|---|---|---|---|
| MCM:HO2CO35C7 $C_7H_{12}O_3$ LULWLYVLBHVZLB-UHFFFAOYSA-N | $3.5\times10^3$ $1.1\times10^4$ $5.6\times10^1$ | | Wang et al. (2017) Wang et al. (2017) Wang et al. (2017) | Q Q Q | 80, 238 80, 239 80, 240 |
| MCM:HO2CO5C7 $C_7H_{14}O_2$ DWMNUVZLRBCGOL-UHFFFAOYSA-N | $8.7\times10^1$ $1.3\times10^3$ $2.1\times10^2$ | | Wang et al. (2017) Wang et al. (2017) Wang et al. (2017) | Q Q Q | 80, 238 80, 239 80, 240 |
| MCM:HO3CO6C7 $C_7H_{14}O_2$ MQRALIJAASQPNT-UHFFFAOYSA-N | $2.1\times10^3$ $8.7\times10^1$ $1.6\times10^3$ $1.7\times10^2$ | 11000 | Wieser et al. (2023) Wang et al. (2017) Wang et al. (2017) Wang et al. (2017) | Q Q Q Q | 437 80, 238 80, 239 80, 240 |
| MCM:PTLQCO $C_7H_6O_4$ QCLUMLMJJNMFFX-UHFFFAOYSA-N | $5.6\times10^6$ $6.2\times10^9$ $6.0\times10^5$ | | Wang et al. (2017) Wang et al. (2017) Wang et al. (2017) | Q Q Q | 80, 238 80, 239 80, 240 |
| MCM:PTLQOH $C_7H_8O_4$ FWCZASAULMADEU-UHFFFAOYSA-N | $2.4\times10^6$ $5.1\times10^8$ $1.2\times10^5$ | | Wang et al. (2017) Wang et al. (2017) Wang et al. (2017) | Q Q Q | 80, 238 80, 239 80, 240 |
| MCM:PTLQOOH $C_7H_8O_5$ ULCYCDVSDDOIBS-UHFFFAOYSA-N | $3.6\times10^9$ $1.8\times10^9$ $3.8\times10^5$ | | Wang et al. (2017) Wang et al. (2017) Wang et al. (2017) | Q Q Q | 80, 238 80, 239 80, 240 |
| MCM:TLCOBIPEOH $C_7H_8O_4$ BTRXIMYXFXXDNL-UHFFFAOYSA-N | $3.3\times10^6$ $7.3\times10^2$ $2.3\times10^5$ | | Wang et al. (2017) Wang et al. (2017) Wang et al. (2017) | Q Q Q | 80, 238 80, 239 80, 240 |
| MCM:TLOBIPEROH $C_7H_8O_4$ OGIZZGFCOPEREO-UHFFFAOYSA-N | $3.3\times10^6$ $1.3\times10^3$ $1.1\times10^5$ | | Wang et al. (2017) Wang et al. (2017) Wang et al. (2017) | Q Q Q | 80, 238 80, 239 80, 240 |
| MCM:C6MOCOCO3H $C_8H_{10}O_6$ UAMJQIWVNIVSAA-UHFFFAOYSA-N | $2.1\times10^9$ $1.7\times10^9$ $4.1\times10^1$ | | Wang et al. (2017) Wang et al. (2017) Wang et al. (2017) | Q Q Q | 80, 238 80, 239 80, 240 |
| MCM:C7CO2M5OH $C_8H_{12}O_3$ VIGCXZSBZQABDL-UHFFFAOYSA-N | $7.8\times10^3$ $2.0\times10^4$ $2.8\times10^3$ | | Wang et al. (2017) Wang et al. (2017) Wang et al. (2017) | Q Q Q | 80, 238 80, 239 80, 240 |
| MCM:C7CO2OCO3H $C_8H_{10}O_6$ PYNUJNZYCLWJPS-UHFFFAOYSA-N | $2.1\times10^9$ $2.0\times10^9$ $4.9\times10^1$ | | Wang et al. (2017) Wang et al. (2017) Wang et al. (2017) | Q Q Q | 80, 238 80, 239 80, 240 |
| MCM:C7M2CO5OH $C_8H_{12}O_3$ IWFVDGADKJRMAI-UHFFFAOYSA-N | $9.1\times10^3$ $8.5\times10^4$ $2.5\times10^3$ | | Wang et al. (2017) Wang et al. (2017) Wang et al. (2017) | Q Q Q | 80, 238 80, 239 80, 240 |
| MCM:C817OH $C_8H_{14}O_3$ HXKXVESDDNIBIQ-UHFFFAOYSA-N | $5.4\times10^4$ $4.9\times10^5$ $2.5\times10^4$ | | Wang et al. (2017) Wang et al. (2017) Wang et al. (2017) | Q Q Q | 80, 238 80, 239 80, 240 |



Table A3.6: Ketones (RCOR) (...continued)

| Substance Formula (Trivial Name) [CAS Registry Number] InChIKey | $H_s^{cp}$ (at $T^{\ominus}$) $\left[ \dfrac{\text{mol}}{\text{m}^3\,\text{Pa}} \right]$ | $\dfrac{\text{d}\ln H_s^{cp}}{\text{d}(1/T)}$ [K] | Reference | Type | Note |
|---|---|---|---|---|---|
| MCM:C818CO $C_8H_{12}O_4$ CRQAGKNTHDNGEW-UHFFFAOYSA-N | $6.5\times10^6$ $3.6\times10^7$ $1.7\times10^7$ $1.3\times10^3$ | 16000 | Wieser et al. (2023) Wang et al. (2017) Wang et al. (2017) Wang et al. (2017) | Q Q Q Q | 437 80, 238 80, 239 80, 240 |
| MCM:C818OH $C_8H_{14}O_4$ GYBDPSWJEGVQPC-UHFFFAOYSA-N | $8.0\times10^6$ $2.9\times10^8$ $7.4\times10^4$ | | Wang et al. (2017) Wang et al. (2017) Wang et al. (2017) | Q Q Q | 80, 238 80, 239 80, 240 |
| MCM:C818OOH $C_8H_{14}O_5$ DLMBPALZQHFDOH-UHFFFAOYSA-N | $5.7\times10^8$ $5.0\times10^9$ $1.0\times10^8$ $9.1\times10^4$ | 17000 | Wieser et al. (2023) Wang et al. (2017) Wang et al. (2017) Wang et al. (2017) | Q Q Q Q | 437 80, 238 80, 239 80, 240 |
| MCM:C819OOH $C_8H_{14}O_5$ JRXMSSDNDAZISI-UHFFFAOYSA-N | $2.1\times10^7$ $3.5\times10^9$ $6.3\times10^7$ $6.3\times10^4$ | 17000 | Wieser et al. (2023) Wang et al. (2017) Wang et al. (2017) Wang et al. (2017) | Q Q Q Q | 437 80, 238 80, 239 80, 240 |
| MCM:C81OOH $C_8H_{16}O_4$ SFFBCYBCCUXGLX-UHFFFAOYSA-N | $6.8\times10^6$ $3.3\times10^5$ $1.3\times10^4$ | | Wang et al. (2017) Wang et al. (2017) Wang et al. (2017) | Q Q Q | 80, 238 80, 239 80, 240 |
| MCM:C820OOH $C_8H_{12}O_6$ MPCACGHSMGOFNV-UHFFFAOYSA-N | $2.3\times10^{12}$ $1.4\times10^9$ $1.6\times10^4$ | | Wang et al. (2017) Wang et al. (2017) Wang et al. (2017) | Q Q Q | 80, 238 80, 239 80, 240 |
| MCM:C826CO3OH $C_8H_{12}O_3$ GGSZUEZFMJTLBX-UHFFFAOYSA-N | $1.1\times10^4$ $5.6\times10^4$ $2.0\times10^3$ | | Wang et al. (2017) Wang et al. (2017) Wang et al. (2017) | Q Q Q | 80, 238 80, 239 80, 240 |
| MCM:C827OH $C_8H_{14}O_3$ VYHQZXLAUHJBAB-UHFFFAOYSA-N | $1.9\times10^3$ $5.4\times10^4$ $1.4\times10^3$ | | Wang et al. (2017) Wang et al. (2017) Wang et al. (2017) | Q Q Q | 80, 238 80, 239 80, 240 |
| MCM:C829OH $C_8H_{14}O_4$ ZFRREEXTYBESJX-UHFFFAOYSA-N | $1.0\times10^8$ $1.0\times10^8$ $1.4\times10^4$ | | Wang et al. (2017) Wang et al. (2017) Wang et al. (2017) | Q Q Q | 80, 238 80, 239 80, 240 |
| MCM:C829OOH $C_8H_{14}O_5$ PFFZQAOPEABNHW-UHFFFAOYSA-N | $2.8\times10^9$ $8.0\times10^7$ $9.3\times10^3$ | | Wang et al. (2017) Wang et al. (2017) Wang et al. (2017) | Q Q Q | 80, 238 80, 239 80, 240 |
| MCM:C82OOH $C_8H_{16}O_4$ AQECYOJQUIJAFS-UHFFFAOYSA-N | $8.5\times10^6$ $6.8\times10^5$ $1.8\times10^3$ | | Wang et al. (2017) Wang et al. (2017) Wang et al. (2017) | Q Q Q | 80, 238 80, 239 80, 240 |
| MCM:C84OH $C_8H_{14}O_3$ HXKSWDLNZPICFQ-UHFFFAOYSA-N | $2.8\times10^3$ $4.1\times10^4$ $5.0\times10^2$ | | Wang et al. (2017) Wang et al. (2017) Wang et al. (2017) | Q Q Q | 80, 238 80, 239 80, 240 |





Table A3.6: Ketones (RCOR) (...continued)

| Substance Formula (Trivial Name) [CAS Registry Number] InChIKey | $H_s^{cp}$ (at $T^{\ominus}$) $\left[\dfrac{\mathrm{mol}}{\mathrm{m^3\,Pa}}\right]$ | $\dfrac{\mathrm{d}\ln H_s^{cp}}{\mathrm{d}(1/T)}$ [K] | Reference | Type | Note |
|---|---|---|---|---|---|
| MCM:C88OH | $6.0\times10^3$ | | Wang et al. (2017) | Q | 80, 238 |
| $C_8H_{12}O_3$ | $1.1\times10^6$ | | Wang et al. (2017) | Q | 80, 239 |
| DCTATDQRTQNXHW-UHFFFAOYSA-N | $3.9\times10^3$ | | Wang et al. (2017) | Q | 80, 240 |
| MCM:EBZOBPEROH | $2.8\times10^6$ | | Wang et al. (2017) | Q | 80, 238 |
| $C_8H_{10}O_4$ | $8.3\times10^2$ | | Wang et al. (2017) | Q | 80, 239 |
| YDVGTQTYIBCIQC-UHFFFAOYSA-N | $6.0\times10^5$ | | Wang et al. (2017) | Q | 80, 240 |
| MCM:HO34CO6C8 | $1.0\times10^5$ | | Wang et al. (2017) | Q | 80, 238 |
| $C_8H_{16}O_3$ | $1.7\times10^5$ | | Wang et al. (2017) | Q | 80, 239 |
| GQDVWZAGWHUXKX-UHFFFAOYSA-N | $3.3\times10^2$ | | Wang et al. (2017) | Q | 80, 240 |
| MCM:HO3CO46C8 | $2.8\times10^3$ | | Wang et al. (2017) | Q | 80, 238 |
| $C_8H_{14}O_3$ | $8.3\times10^3$ | | Wang et al. (2017) | Q | 80, 239 |
| LLYUJOAMIUBKSA-UHFFFAOYSA-N | $3.0\times10^1$ | | Wang et al. (2017) | Q | 80, 240 |
| MCM:HO3CO6C8 | $8.0\times10^1$ | | Wang et al. (2017) | Q | 80, 238 |
| $C_8H_{16}O_2$ | $9.6\times10^2$ | | Wang et al. (2017) | Q | 80, 239 |
| VJDZJBNTVLGPEM-UHFFFAOYSA-N | $8.7\times10^1$ | | Wang et al. (2017) | Q | 80, 240 |
| MCM:MXYOBPEROH | $1.9\times10^6$ | | Wang et al. (2017) | Q | 80, 238 |
| $C_8H_{10}O_4$ | $3.0\times10^2$ | | Wang et al. (2017) | Q | 80, 239 |
| BQJOHFWATKFGAQ-UHFFFAOYSA-N | $4.5\times10^4$ | | Wang et al. (2017) | Q | 80, 240 |
| MCM:MXYQOH | $1.3\times10^6$ | | Wang et al. (2017) | Q | 80, 238 |
| $C_8H_{10}O_4$ | $3.0\times10^8$ | | Wang et al. (2017) | Q | 80, 239 |
| DNSXSYZLXRHYTM-UHFFFAOYSA-N | $3.5\times10^4$ | | Wang et al. (2017) | Q | 80, 240 |
| MCM:MXYQOOH | $1.9\times10^9$ | | Wang et al. (2017) | Q | 80, 238 |
| $C_8H_{10}O_5$ | $3.0\times10^8$ | | Wang et al. (2017) | Q | 80, 239 |
| UAWVAAITGKOKNW-UHFFFAOYSA-N | $5.9\times10^7$ | | Wang et al. (2017) | Q | 80, 240 |
| MCM:OXYQCO | $3.4\times10^6$ | | Wang et al. (2017) | Q | 80, 238 |
| $C_8H_8O_4$ | $6.2\times10^9$ | | Wang et al. (2017) | Q | 80, 239 |
| QSCBEAAIZXLBQA-UHFFFAOYSA-N | $1.8\times10^5$ | | Wang et al. (2017) | Q | 80, 240 |
| MCM:OXYQOH | $1.6\times10^6$ | | Wang et al. (2017) | Q | 80, 238 |
| $C_8H_{10}O_4$ | $3.5\times10^8$ | | Wang et al. (2017) | Q | 80, 239 |
| LJMMKZLQFVIDLQ-UHFFFAOYSA-N | $7.6\times10^4$ | | Wang et al. (2017) | Q | 80, 240 |
| MCM:OXYQOOH | $2.3\times10^9$ | | Wang et al. (2017) | Q | 80, 238 |
| $C_8H_{10}O_5$ | $1.2\times10^9$ | | Wang et al. (2017) | Q | 80, 239 |
| NOWQTCYBLCRAEJ-UHFFFAOYSA-N | $1.4\times10^5$ | | Wang et al. (2017) | Q | 80, 240 |
| MCM:PXYQOH | $1.3\times10^6$ | | Wang et al. (2017) | Q | 80, 238 |
| $C_8H_{10}O_4$ | $3.0\times10^8$ | | Wang et al. (2017) | Q | 80, 239 |
| RORXCXPFLRGHQV-UHFFFAOYSA-N | $5.8\times10^4$ | | Wang et al. (2017) | Q | 80, 240 |
| MCM:PXYQOOH | $1.9\times10^9$ | | Wang et al. (2017) | Q | 80, 238 |
| $C_8H_{10}O_5$ | $4.2\times10^8$ | | Wang et al. (2017) | Q | 80, 239 |
| FYDHRLZPZZTRLJ-UHFFFAOYSA-N | $2.4\times10^6$ | | Wang et al. (2017) | Q | 80, 240 |



Table A3.6: Ketones (RCOR) (. . . continued)

| Substance Formula (Trivial Name) [CAS Registry Number] InChIKey | $H_s^{cp}$ (at $T^{\ominus}$) $\left[\dfrac{\mathrm{mol}}{\mathrm{m}^3\,\mathrm{Pa}}\right]$ | $\dfrac{\mathrm{d}\ln H_s^{cp}}{\mathrm{d}(1/T)}$ [K] | Reference | Type | Note |
|---|---|---|---|---|---|
| MCM:C6EO2HCO3H $C_9H_{12}O_6$ KMUWHHLOUXRZEC-UHFFFAOYSA-N | $1.7\times10^9$ $1.6\times10^9$ $3.1\times10^1$ | | Wang et al. (2017) Wang et al. (2017) Wang et al. (2017) | Q Q Q | 80, 238 80, 239 80, 240 |
| MCM:C7MJPCO3H $C_9H_{12}O_6$ HVGFJXMJLWNDQT-UHFFFAOYSA-N | $1.7\times10^9$ $1.2\times10^9$ $6.2\times10^1$ | | Wang et al. (2017) Wang et al. (2017) Wang et al. (2017) | Q Q Q | 80, 238 80, 239 80, 240 |
| MCM:C7MOCOCO3H $C_9H_{12}O_6$ UXMGGYROBLVMKA-UHFFFAOYSA-N | $1.4\times10^9$ $2.3\times10^9$ $9.8\times10^1$ | | Wang et al. (2017) Wang et al. (2017) Wang et al. (2017) | Q Q Q | 80, 238 80, 239 80, 240 |
| MCM:C8M2CO6OH $C_9H_{14}O_3$ AUYUFIFJEJTOMG-UHFFFAOYSA-N | $7.6\times10^3$ $4.8\times10^4$ $8.5\times10^2$ | | Wang et al. (2017) Wang et al. (2017) Wang et al. (2017) | Q Q Q | 80, 238 80, 239 80, 240 |
| MCM:C917OH $C_9H_{14}O_3$ DVAQWOBQFULNSX-UHFFFAOYSA-N | $9.8\times10^4$ $4.3\times10^6$ $3.1\times10^5$ | | Wang et al. (2017) Wang et al. (2017) Wang et al. (2017) | Q Q Q | 80, 238 80, 239 80, 240 |
| MCM:C920OOH $C_9H_{16}O_4$ CXKZGZZGALDRTB-UHFFFAOYSA-N | $6.5\times10^5$ $1.6\times10^6$ $6.9\times10^5$ | | Wang et al. (2017) Wang et al. (2017) Wang et al. (2017) | Q Q Q | 80, 238 80, 239 80, 240 |
| MCM:C921OOH $C_9H_{16}O_5$ MIFQELLQXZRDIQ-UHFFFAOYSA-N | $1.2\times10^9$ $3.2\times10^9$ $7.1\times10^7$ | | Wang et al. (2017) Wang et al. (2017) Wang et al. (2017) | Q Q Q | 80, 238 80, 239 80, 240 |
| MCM:C922OOH $C_9H_{16}O_6$ IOTRHTNOHNKLJT-UHFFFAOYSA-N | $3.6\times10^{11}$ $7.8\times10^9$ $6.6\times10^6$ | | Wang et al. (2017) Wang et al. (2017) Wang et al. (2017) | Q Q Q | 80, 238 80, 239 80, 240 |
| MCM:C923OH $C_9H_{16}O_2$ VZYSLJNPEGILSI-UHFFFAOYSA-N | $5.3\times10^3$ $1.5\times10^2$ $2.0\times10^3$ $7.4\times10^2$ | 12000 | Wieser et al. (2023) Wang et al. (2017) Wang et al. (2017) Wang et al. (2017) | Q Q Q Q | 437 80, 238 80, 239 80, 240 |
| MCM:C924CO $C_9H_{14}O_3$ IEIWFPIZSBHOOD-UHFFFAOYSA-N | $1.0\times10^5$ $5.6\times10^4$ $4.4\times10^2$ | | Wang et al. (2017) Wang et al. (2017) Wang et al. (2017) | Q Q Q | 80, 238 80, 239 80, 240 |
| MCM:C924OH $C_9H_{16}O_3$ OXSRPARPRXNCEH-UHFFFAOYSA-N | $3.8\times10^5$ $2.2\times10^4$ $1.4\times10^5$ $2.7\times10^3$ | 16000 | Wieser et al. (2023) Wang et al. (2017) Wang et al. (2017) Wang et al. (2017) | Q Q Q Q | 437 80, 238 80, 239 80, 240 |
| MCM:C924OOH $C_9H_{16}O_4$ BVODBOGLMHSHGK-UHFFFAOYSA-N | $3.0\times10^6$ $1.2\times10^7$ $3.1\times10^5$ $5.0\times10^4$ | 16000 | Wieser et al. (2023) Wang et al. (2017) Wang et al. (2017) Wang et al. (2017) | Q Q Q Q | 437 80, 238 80, 239 80, 240 |





Table A3.6: Ketones (RCOR) (... continued)

| Substance<br>Formula<br>(Trivial Name)<br>[CAS Registry Number]<br>InChIKey | $H_s^{cp}$<br>(at $T^\ominus$)<br>$\left[\dfrac{\text{mol}}{\text{m}^3\,\text{Pa}}\right]$ | $\dfrac{\text{d}\ln H_s^{cp}}{\text{d}(1/T)}$<br><br>[K] | Reference | Type | Note |
|---|---|---|---|---|---|
| MCM:C925OOH | $9.1\times10^{12}$ | | Wang et al. (2017) | Q | 80, 238 |
| $C_9H_{16}O_6$ | $5.0\times10^{10}$ | | Wang et al. (2017) | Q | 80, 239 |
| BGVLBDREANINMN-UHFFFAOYSA-N | $1.6\times10^{7}$ | | Wang et al. (2017) | Q | 80, 240 |
| MCM:C927OH | $2.9\times10^{5}$ | | Wang et al. (2017) | Q | 80, 238 |
| $C_9H_{16}O_3$ | $1.2\times10^{7}$ | | Wang et al. (2017) | Q | 80, 239 |
| JILOVKBSEAXPMT-UHFFFAOYSA-N | $9.3\times10^{4}$ | | Wang et al. (2017) | Q | 80, 240 |
| MCM:C927OOH | $7.6\times10^{6}$ | | Wang et al. (2017) | Q | 80, 238 |
| $C_9H_{16}O_4$ | $8.9\times10^{6}$ | | Wang et al. (2017) | Q | 80, 239 |
| UVVCFFCMOUIWEJ-UHFFFAOYSA-N | $6.6\times10^{4}$ | | Wang et al. (2017) | Q | 80, 240 |
| MCM:C928OH | $2.9\times10^{4}$ | | Wang et al. (2017) | Q | 80, 238 |
| $C_9H_{16}O_3$ | $3.8\times10^{5}$ | | Wang et al. (2017) | Q | 80, 239 |
| QGCNXROUFNWXOO-UHFFFAOYSA-N | $1.6\times10^{3}$ | | Wang et al. (2017) | Q | 80, 240 |
| MCM:C929CO | $2.0\times10^{7}$ | | Wang et al. (2017) | Q | 80, 238 |
| $C_9H_{14}O_4$ | $5.4\times10^{6}$ | | Wang et al. (2017) | Q | 80, 239 |
| BWILLNTUIBCKIM-UHFFFAOYSA-N | $3.5\times10^{2}$ | | Wang et al. (2017) | Q | 80, 240 |
| MCM:C929OH | $4.4\times10^{6}$ | | Wang et al. (2017) | Q | 80, 238 |
| $C_9H_{16}O_4$ | $5.9\times10^{8}$ | | Wang et al. (2017) | Q | 80, 239 |
| PMWQIXWFMSCXRH-UHFFFAOYSA-N | $6.9\times10^{3}$ | | Wang et al. (2017) | Q | 80, 240 |
| MCM:C929OOH | $2.9\times10^{9}$ | | Wang et al. (2017) | Q | 80, 238 |
| $C_9H_{16}O_5$ | $3.3\times10^{8}$ | | Wang et al. (2017) | Q | 80, 239 |
| IKUORCWWMOEDGL-UHFFFAOYSA-N | $1.2\times10^{5}$ | | Wang et al. (2017) | Q | 80, 240 |
| MCM:C92OOH | $5.3\times10^{6}$ | | Wang et al. (2017) | Q | 80, 238 |
| $C_9H_{18}O_4$ | $2.9\times10^{5}$ | | Wang et al. (2017) | Q | 80, 239 |
| MOTPUWPPGZYDTM-UHFFFAOYSA-N | $2.2\times10^{4}$ | | Wang et al. (2017) | Q | 80, 240 |
| MCM:C93CO | $2.2\times10^{3}$ | | Wang et al. (2017) | Q | 80, 238 |
| $C_9H_{16}O_3$ | $6.6\times10^{3}$ | | Wang et al. (2017) | Q | 80, 239 |
| NCWLHDSPZIENNF-UHFFFAOYSA-N | $2.8\times10^{1}$ | | Wang et al. (2017) | Q | 80, 240 |
| MCM:C93OH | $8.0\times10^{4}$ | | Wang et al. (2017) | Q | 80, 238 |
| $C_9H_{18}O_3$ | $1.5\times10^{5}$ | | Wang et al. (2017) | Q | 80, 239 |
| XOZZRVPGLURXSY-UHFFFAOYSA-N | $2.7\times10^{2}$ | | Wang et al. (2017) | Q | 80, 240 |
| MCM:C93OOH | $6.8\times10^{6}$ | | Wang et al. (2017) | Q | 80, 238 |
| $C_9H_{18}O_4$ | $3.3\times10^{5}$ | | Wang et al. (2017) | Q | 80, 239 |
| DSWUVSMVGXSNEV-UHFFFAOYSA-N | $8.0\times10^{3}$ | | Wang et al. (2017) | Q | 80, 240 |
| MCM:C94OH | $2.2\times10^{3}$ | | Wang et al. (2017) | Q | 80, 238 |
| $C_9H_{16}O_3$ | $2.6\times10^{4}$ | | Wang et al. (2017) | Q | 80, 239 |
| RCWXDPYULCSTKX-UHFFFAOYSA-N | $3.6\times10^{2}$ | | Wang et al. (2017) | Q | 80, 240 |
| MCM:C96OH | $1.7\times10^{2}$ | | Wang et al. (2017) | Q | 80, 238 |
| $C_9H_{16}O_2$ | $3.3\times10^{3}$ | | Wang et al. (2017) | Q | 80, 239 |
| RRFQMYCDTHUGSU-UHFFFAOYSA-N | $1.5\times10^{4}$ | | Wang et al. (2017) | Q | 80, 240 |



Table A3.6: Ketones (RCOR) (...continued)

| Substance Formula (Trivial Name) [CAS Registry Number] InChIKey | $H_s^{cp}$ (at $T^\ominus$) $\left[\dfrac{\text{mol}}{\text{m}^3\,\text{Pa}}\right]$ | $\dfrac{\text{d}\ln H_s^{cp}}{\text{d}(1/T)}$ [K] | Reference | Type | Note |
|---|---|---|---|---|---|
| MCM:C97OH | $1.4\times10^4$ | | Wang et al. (2017) | Q | 80, 238 |
| $C_9H_{16}O_3$ | $1.2\times10^6$ | | Wang et al. (2017) | Q | 80, 239 |
| QWYKJUJQBVIHQW-UHFFFAOYSA-N | $8.0\times10^4$ | | Wang et al. (2017) | Q | 80, 240 |
| MCM:C97OOH | $8.1\times10^6$ | | Wang et al. (2017) | Q | 80, 238 |
| $C_9H_{16}O_4$ | $3.9\times10^5$ | | Wang et al. (2017) | Q | 80, 239 |
| AXRJNDGSGFVBSX-UHFFFAOYSA-N | $1.9\times10^6$ | | Wang et al. (2017) | Q | 80, 240 |
| MCM:C98OH | $9.3\times10^7$ | | Wang et al. (2017) | Q | 80, 238 |
| $C_9H_{16}O_4$ | $1.0\times10^8$ | | Wang et al. (2017) | Q | 80, 239 |
| BWKWRXKCGJAJAO-UHFFFAOYSA-N | $4.2\times10^4$ | | Wang et al. (2017) | Q | 80, 240 |
| MCM:C98OOH | $2.9\times10^9$ | | Wang et al. (2017) | Q | 80, 238 |
| $C_9H_{16}O_5$ | $3.3\times10^7$ | | Wang et al. (2017) | Q | 80, 239 |
| MEUGYCLQCHCPCY-UHFFFAOYSA-N | $1.1\times10^5$ | | Wang et al. (2017) | Q | 80, 240 |
| MCM:C9DCOH | $4.0\times10^5$ | | Wang et al. (2017) | Q | 80, 238 |
| $C_9H_{12}O_3$ | $1.9\times10^7$ | | Wang et al. (2017) | Q | 80, 239 |
| DIFRJFLUSYQBJM-UHFFFAOYSA-N | $4.9\times10^6$ | | Wang et al. (2017) | Q | 80, 240 |
| MCM:HO4CO7C9 | $6.3\times10^1$ | | Wang et al. (2017) | Q | 80, 238 |
| $C_9H_{18}O_2$ | $8.3\times10^2$ | | Wang et al. (2017) | Q | 80, 239 |
| BPRAKHMXALGYKO-UHFFFAOYSA-N | $7.1\times10^1$ | | Wang et al. (2017) | Q | 80, 240 |
| MCM:IPBZOBPROH | $2.6\times10^6$ | | Wang et al. (2017) | Q | 80, 238 |
| $C_9H_{12}O_4$ | $7.1\times10^2$ | | Wang et al. (2017) | Q | 80, 239 |
| WRQNGVASMLSOGA-UHFFFAOYSA-N | $3.6\times10^4$ | | Wang et al. (2017) | Q | 80, 240 |
| MCM:IPRBQCO | $4.1\times10^6$ | | Wang et al. (2017) | Q | 80, 238 |
| $C_9H_{10}O_4$ | $2.6\times10^9$ | | Wang et al. (2017) | Q | 80, 239 |
| UYSSVGYDANWBFJ-UHFFFAOYSA-N | $1.2\times10^5$ | | Wang et al. (2017) | Q | 80, 240 |
| MCM:IPRBQOH | $1.8\times10^6$ | | Wang et al. (2017) | Q | 80, 238 |
| $C_9H_{12}O_4$ | $1.7\times10^8$ | | Wang et al. (2017) | Q | 80, 239 |
| BQKYQBQGKYWJSQ-UHFFFAOYSA-N | $7.6\times10^4$ | | Wang et al. (2017) | Q | 80, 240 |
| MCM:IPRBQOOH | $2.9\times10^9$ | | Wang et al. (2017) | Q | 80, 238 |
| $C_9H_{12}O_5$ | $6.9\times10^8$ | | Wang et al. (2017) | Q | 80, 239 |
| CCPBRYDTGCPJNX-UHFFFAOYSA-N | $1.2\times10^5$ | | Wang et al. (2017) | Q | 80, 240 |
| MCM:LMKAOH | $2.0\times10^5$ | | Wang et al. (2017) | Q | 80, 238 |
| $C_9H_{16}O_3$ | $6.2\times10^6$ | | Wang et al. (2017) | Q | 80, 239 |
| RDFFVUMNEKCADL-UHFFFAOYSA-N | $1.2\times10^6$ | | Wang et al. (2017) | Q | 80, 240 |
| MCM:LMKAOOH | $1.5\times10^7$ | | Wang et al. (2017) | Q | 80, 238 |
| $C_9H_{16}O_4$ | $2.7\times10^7$ | | Wang et al. (2017) | Q | 80, 239 |
| ZYSXBTHWQDNLBX-UHFFFAOYSA-N | $3.6\times10^6$ | | Wang et al. (2017) | Q | 80, 240 |
| MCM:LMKBCO | $5.0\times10^3$ | | Wang et al. (2017) | Q | 80, 238 |
| $C_9H_{14}O_3$ | $3.7\times10^5$ | | Wang et al. (2017) | Q | 80, 239 |
| QJYRJOZGIVJKQK-UHFFFAOYSA-N | $2.9\times10^4$ | | Wang et al. (2017) | Q | 80, 240 |





Table A3.6: Ketones (RCOR) (...continued)

| Substance<br>Formula<br>(Trivial Name)<br>[CAS Registry Number]<br>InChIKey | $H_s^{cp}$<br>(at $T^{\ominus}$)<br>$\left[\dfrac{\mathrm{mol}}{\mathrm{m^3\,Pa}}\right]$ | $\dfrac{\mathrm{d}\ln H_s^{cp}}{\mathrm{d}(1/T)}$<br><br>[K] | Reference | Type | Note |
|---|---|---|---|---|---|
| MCM:LMKBOOH | $1.5\times10^7$ | | Wang et al. (2017) | Q | 80, 238 |
| $C_9H_{16}O_4$ | $2.6\times10^7$ | | Wang et al. (2017) | Q | 80, 239 |
| PMLHMEIQMSUJKR-UHFFFAOYSA-N | $8.9\times10^6$ | | Wang et al. (2017) | Q | 80, 240 |
| MCM:METLOBPROH | $1.5\times10^6$ | | Wang et al. (2017) | Q | 80, 238 |
| $C_9H_{12}O_4$ | $1.8\times10^2$ | | Wang et al. (2017) | Q | 80, 239 |
| ZSTYNOBMCJBFRA-UHFFFAOYSA-N | $2.5\times10^5$ | | Wang et al. (2017) | Q | 80, 240 |
| MCM:METLQOH | $1.1\times10^6$ | | Wang et al. (2017) | Q | 80, 238 |
| $C_9H_{12}O_4$ | $2.0\times10^8$ | | Wang et al. (2017) | Q | 80, 239 |
| YIUSRNITCRMQJW-UHFFFAOYSA-N | $2.1\times10^4$ | | Wang et al. (2017) | Q | 80, 240 |
| MCM:METLQOOH | $1.7\times10^9$ | | Wang et al. (2017) | Q | 80, 238 |
| $C_9H_{12}O_5$ | $1.3\times10^8$ | | Wang et al. (2017) | Q | 80, 239 |
| JBNOKZJMGKNXDA-UHFFFAOYSA-N | $5.6\times10^4$ | | Wang et al. (2017) | Q | 80, 240 |
| MCM:NOPINAOH | $5.4\times10^2$ | | Wang et al. (2017) | Q | 80, 238 |
| $C_9H_{14}O_2$ | $2.5\times10^4$ | | Wang et al. (2017) | Q | 80, 239 |
| CVWGOJDPALGYKB-UHFFFAOYSA-N | $4.0\times10^4$ | | Wang et al. (2017) | Q | 80, 240 |
| MCM:NOPINBOH | $5.4\times10^2$ | | Wang et al. (2017) | Q | 80, 238 |
| $C_9H_{14}O_2$ | $3.2\times10^4$ | | Wang et al. (2017) | Q | 80, 239 |
| FPBMXOVOMRHDTB-UHFFFAOYSA-N | $2.7\times10^4$ | | Wang et al. (2017) | Q | 80, 240 |
| MCM:NOPINCOH | $3.1\times10^2$ | | Wang et al. (2017) | Q | 80, 238 |
| $C_9H_{14}O_2$ | $9.3\times10^3$ | | Wang et al. (2017) | Q | 80, 239 |
| OXRZVROAQQDWAA-UHFFFAOYSA-N | $3.2\times10^4$ | | Wang et al. (2017) | Q | 80, 240 |
| MCM:NOPINDOH | $2.8\times10^1$ | | Wang et al. (2017) | Q | 80, 238 |
| $C_9H_{14}O_2$ | $7.3\times10^2$ | | Wang et al. (2017) | Q | 80, 239 |
| MLOBETUCFXAOOL-UHFFFAOYSA-N | $8.5\times10^1$ | | Wang et al. (2017) | Q | 80, 240 |
| MCM:OETLQCO | $3.0\times10^6$ | | Wang et al. (2017) | Q | 80, 238 |
| $C_9H_{10}O_4$ | $4.7\times10^9$ | | Wang et al. (2017) | Q | 80, 239 |
| WOEKXHYIEUECKR-UHFFFAOYSA-N | $1.0\times10^5$ | | Wang et al. (2017) | Q | 80, 240 |
| MCM:OETLQOH | $1.3\times10^6$ | | Wang et al. (2017) | Q | 80, 238 |
| $C_9H_{12}O_4$ | $2.0\times10^8$ | | Wang et al. (2017) | Q | 80, 239 |
| DZHQWTGZQAKVBN-UHFFFAOYSA-N | $5.9\times10^4$ | | Wang et al. (2017) | Q | 80, 240 |
| MCM:OETLQOOH | $1.9\times10^9$ | | Wang et al. (2017) | Q | 80, 238 |
| $C_9H_{12}O_5$ | $8.0\times10^8$ | | Wang et al. (2017) | Q | 80, 239 |
| KKWSBZSFLIKUTL-UHFFFAOYSA-N | $2.1\times10^5$ | | Wang et al. (2017) | Q | 80, 240 |
| MCM:PBZOBPEROH | $2.1\times10^6$ | | Wang et al. (2017) | Q | 80, 238 |
| $C_9H_{12}O_4$ | $5.5\times10^2$ | | Wang et al. (2017) | Q | 80, 239 |
| JGNLEXNHGWFGHL-UHFFFAOYSA-N | $5.9\times10^4$ | | Wang et al. (2017) | Q | 80, 240 |
| MCM:PETLQOH | $1.1\times10^6$ | | Wang et al. (2017) | Q | 80, 238 |
| $C_9H_{12}O_4$ | $2.2\times10^8$ | | Wang et al. (2017) | Q | 80, 239 |
| KZOAJFYPECZGSC-UHFFFAOYSA-N | $1.1\times10^4$ | | Wang et al. (2017) | Q | 80, 240 |



Table A3.6: Ketones (RCOR) (...continued)

| Substance Formula (Trivial Name) [CAS Registry Number] InChIKey | $H_s^{cp}$ (at $T^{\ominus}$) $\left[\dfrac{\mathrm{mol}}{\mathrm{m}^3\,\mathrm{Pa}}\right]$ | $\dfrac{\mathrm{d}\ln H_s^{cp}}{\mathrm{d}(1/T)}$ [K] | Reference | Type | Note |
|---|---|---|---|---|---|
| MCM:PETLQOOH $C_9H_{12}O_5$ BJRXYYOXPIRVAX-UHFFFAOYSA-N | $1.7\times10^9$ $2.0\times10^8$ $2.0\times10^5$ | | Wang et al. (2017) Wang et al. (2017) Wang et al. (2017) | Q Q Q | 80, 238 80, 239 80, 240 |
| MCM:TM123OBPOH $C_9H_{12}O_4$ WQSRSICCLTUNRX-UHFFFAOYSA-N | $1.0\times10^6$ $1.7\times10^2$ $4.5\times10^3$ | | Wang et al. (2017) Wang et al. (2017) Wang et al. (2017) | Q Q Q | 80, 238 80, 239 80, 240 |
| MCM:TM124QOH $C_9H_{12}O_4$ GVKLCWOTVYVOTB-UHFFFAOYSA-N | $8.9\times10^5$ $1.7\times10^8$ $2.6\times10^4$ | | Wang et al. (2017) Wang et al. (2017) Wang et al. (2017) | Q Q Q | 80, 238 80, 239 80, 240 |
| MCM:TM124QOOH $C_9H_{12}O_5$ SLEWHRYFRSXVOQ-UHFFFAOYSA-N | $1.3\times10^9$ $2.6\times10^8$ $3.2\times10^7$ | | Wang et al. (2017) Wang et al. (2017) Wang et al. (2017) | Q Q Q | 80, 238 80, 239 80, 240 |
| MCM:TM135OBPOH $C_9H_{12}O_4$ MWFXWFJYOSXFGA-UHFFFAOYSA-N | $1.2\times10^6$ $1.9\times10^2$ $4.2\times10^4$ | | Wang et al. (2017) Wang et al. (2017) Wang et al. (2017) | Q Q Q | 80, 238 80, 239 80, 240 |
| MCM:APINBCO $C_{10}H_{16}O_2$ VZRRCQOUNSHSGB-UHFFFAOYSA-N | $1.5\times10^1$ $4.4\times10^2$ $1.2\times10^2$ | | Wang et al. (2017) Wang et al. (2017) Wang et al. (2017) | Q Q Q | 80, 238 80, 239 80, 240 |
| MCM:C1011OH $C_{10}H_{18}O_2$ FSFKNSSBELMLGT-UHFFFAOYSA-N | $1.3\times10^2$ $2.1\times10^3$ $1.1\times10^2$ | | Wang et al. (2017) Wang et al. (2017) Wang et al. (2017) | Q Q Q | 80, 238 80, 239 80, 240 |
| MCM:C102OOH $C_{10}H_{20}O_4$ QHLMVJFPSLBXKZ-UHFFFAOYSA-N | $4.9\times10^6$ $2.6\times10^5$ $1.1\times10^4$ | | Wang et al. (2017) Wang et al. (2017) Wang et al. (2017) | Q Q Q | 80, 238 80, 239 80, 240 |
| MCM:C103CO $C_{10}H_{18}O_3$ QIIAZEBRWZSMOM-UHFFFAOYSA-N | $2.0\times10^3$ $5.4\times10^3$ $2.5\times10^1$ | | Wang et al. (2017) Wang et al. (2017) Wang et al. (2017) | Q Q Q | 80, 238 80, 239 80, 240 |
| MCM:C103OH $C_{10}H_{20}O_3$ SXEGIXJYQPFAQK-UHFFFAOYSA-N | $7.4\times10^4$ $1.3\times10^5$ $2.5\times10^2$ | | Wang et al. (2017) Wang et al. (2017) Wang et al. (2017) | Q Q Q | 80, 238 80, 239 80, 240 |
| MCM:C103OOH $C_{10}H_{20}O_4$ DOFXWCNTKGZWNU-UHFFFAOYSA-N | $5.5\times10^6$ $3.0\times10^5$ $1.2\times10^4$ | | Wang et al. (2017) Wang et al. (2017) Wang et al. (2017) | Q Q Q | 80, 238 80, 239 80, 240 |
| MCM:C104OH $C_{10}H_{18}O_3$ FHITZQAABUJLCH-UHFFFAOYSA-N | $2.0\times10^3$ $1.9\times10^4$ $4.1\times10^2$ | | Wang et al. (2017) Wang et al. (2017) Wang et al. (2017) | Q Q Q | 80, 238 80, 239 80, 240 |
| MCM:C920CO3H $C_{10}H_{16}O_5$ OFVMBWQGLFFRLM-UHFFFAOYSA-N | $6.6\times10^6$ $9.1\times10^5$ $1.1\times10^4$ | | Wang et al. (2017) Wang et al. (2017) Wang et al. (2017) | Q Q Q | 80, 238 80, 239 80, 240 |





Table A3.6: Ketones (RCOR) (...continued)

| Substance Formula (Trivial Name) [CAS Registry Number] InChIKey | $H_s^{cp}$ (at $T^\ominus$) $\left[\dfrac{\text{mol}}{\text{m}^3\,\text{Pa}}\right]$ | $\dfrac{\text{d}\ln H_s^{cp}}{\text{d}(1/T)}$ [K] | Reference | Type | Note |
|---|---|---|---|---|---|
| MCM:C9M2CO6OH | $5.9\times10^3$ | | Wang et al. (2017) | Q | 80, 238 |
| $C_{10}H_{16}O_3$ | $2.4\times10^4$ | | Wang et al. (2017) | Q | 80, 239 |
| SLPXKTFAPIGCFF-UHFFFAOYSA-N | $4.8\times10^2$ | | Wang et al. (2017) | Q | 80, 240 |
| MCM:DMEOBPROH | $1.0\times10^6$ | | Wang et al. (2017) | Q | 80, 238 |
| $C_{10}H_{14}O_4$ | $1.3\times10^2$ | | Wang et al. (2017) | Q | 80, 239 |
| ZCTKJUASOMJGQC-UHFFFAOYSA-N | $9.8\times10^3$ | | Wang et al. (2017) | Q | 80, 240 |
| MCM:HO5CO8C10 | $5.1\times10^1$ | | Wang et al. (2017) | Q | 80, 238 |
| $C_{10}H_{20}O_2$ | $7.4\times10^2$ | | Wang et al. (2017) | Q | 80, 239 |
| GDOWVFFRHQHNBZ-UHFFFAOYSA-N | $8.3\times10^1$ | | Wang et al. (2017) | Q | 80, 240 |
| MCM:LIMBCO | $1.4\times10^1$ | | Wang et al. (2017) | Q | 80, 238 |
| $C_{10}H_{16}O_2$ | $3.5\times10^2$ | | Wang et al. (2017) | Q | 80, 239 |
| JEQLRDRDFLXSHY-UHFFFAOYSA-N | $1.6\times10^2$ | | Wang et al. (2017) | Q | 80, 240 |
| MCM:C112OOH | $3.8\times10^6$ | | Wang et al. (2017) | Q | 80, 238 |
| $C_{11}H_{22}O_4$ | $2.4\times10^5$ | | Wang et al. (2017) | Q | 80, 239 |
| IFXOTQISYXBGBQ-UHFFFAOYSA-N | $5.4\times10^3$ | | Wang et al. (2017) | Q | 80, 240 |
| MCM:C113CO | $1.6\times10^3$ | | Wang et al. (2017) | Q | 80, 238 |
| $C_{11}H_{20}O_3$ | $4.5\times10^3$ | | Wang et al. (2017) | Q | 80, 239 |
| VJVDGYIYRBYKOQ-UHFFFAOYSA-N | $2.3\times10^1$ | | Wang et al. (2017) | Q | 80, 240 |
| MCM:C113OH | $5.8\times10^4$ | | Wang et al. (2017) | Q | 80, 238 |
| $C_{11}H_{22}O_3$ | $1.1\times10^5$ | | Wang et al. (2017) | Q | 80, 239 |
| IBMABDXBRFZEGO-UHFFFAOYSA-N | $3.6\times10^2$ | | Wang et al. (2017) | Q | 80, 240 |
| MCM:C113OOH | $4.9\times10^6$ | | Wang et al. (2017) | Q | 80, 238 |
| $C_{11}H_{22}O_4$ | $3.0\times10^5$ | | Wang et al. (2017) | Q | 80, 239 |
| PBWNXGFXFCLKSE-UHFFFAOYSA-N | $8.5\times10^2$ | | Wang et al. (2017) | Q | 80, 240 |
| MCM:C114OH | $1.6\times10^3$ | | Wang et al. (2017) | Q | 80, 238 |
| $C_{11}H_{20}O_3$ | $1.4\times10^4$ | | Wang et al. (2017) | Q | 80, 239 |
| UGVQLPIEIICAJP-UHFFFAOYSA-N | $3.7\times10^2$ | | Wang et al. (2017) | Q | 80, 240 |
| MCM:DETLOBPROH | $8.0\times10^5$ | | Wang et al. (2017) | Q | 80, 238 |
| $C_{11}H_{16}O_4$ | $9.1\times10^1$ | | Wang et al. (2017) | Q | 80, 239 |
| QNKVJNYVFVQVHN-UHFFFAOYSA-N | $1.7\times10^4$ | | Wang et al. (2017) | Q | 80, 240 |
| MCM:HO6CO9C11 | $4.6\times10^1$ | | Wang et al. (2017) | Q | 80, 238 |
| $C_{11}H_{22}O_2$ | $6.5\times10^2$ | | Wang et al. (2017) | Q | 80, 239 |
| HEIFQPPUFKWMSE-UHFFFAOYSA-N | $5.8\times10^1$ | | Wang et al. (2017) | Q | 80, 240 |
| MCM:C122OOH | $3.2\times10^6$ | | Wang et al. (2017) | Q | 80, 238 |
| $C_{12}H_{24}O_4$ | $2.3\times10^5$ | | Wang et al. (2017) | Q | 80, 239 |
| PWTYXUGOJRDLPW-UHFFFAOYSA-N | $9.3\times10^3$ | | Wang et al. (2017) | Q | 80, 240 |
| MCM:C123CO | $1.3\times10^3$ | | Wang et al. (2017) | Q | 80, 238 |
| $C_{12}H_{22}O_3$ | $4.1\times10^3$ | | Wang et al. (2017) | Q | 80, 239 |
| IVLDSNPYDYHHKW-UHFFFAOYSA-N | $2.1\times10^1$ | | Wang et al. (2017) | Q | 80, 240 |





Table A3.6: Ketones (RCOR) (...continued)

| Substance Formula (Trivial Name) [CAS Registry Number] InChIKey | $H_s^{cp}$ (at $T^{\ominus}$) $\left[\dfrac{\mathrm{mol}}{\mathrm{m}^3\,\mathrm{Pa}}\right]$ | $\dfrac{\mathrm{d}\ln H_s^{cp}}{\mathrm{d}(1/T)}$ [K] | Reference | Type | Note |
|---|---|---|---|---|---|
| MCM:C123OH | $4.8\times10^4$ | | Wang et al. (2017) | Q | 80, 238 |
| $C_{12}H_{24}O_3$ | $9.6\times10^4$ | | Wang et al. (2017) | Q | 80, 239 |
| FTSJAGYKUWYWCN-UHFFFAOYSA-N | $1.7\times10^2$ | | Wang et al. (2017) | Q | 80, 240 |
| MCM:C123OOH | $4.0\times10^6$ | | Wang et al. (2017) | Q | 80, 238 |
| $C_{12}H_{24}O_4$ | $2.8\times10^5$ | | Wang et al. (2017) | Q | 80, 239 |
| YANZHJXFRFYAEY-UHFFFAOYSA-N | $1.5\times10^4$ | | Wang et al. (2017) | Q | 80, 240 |
| MCM:C124OH | $1.3\times10^3$ | | Wang et al. (2017) | Q | 80, 238 |
| $C_{12}H_{22}O_3$ | $1.3\times10^4$ | | Wang et al. (2017) | Q | 80, 239 |
| ABVYFIKKAQHKSN-UHFFFAOYSA-N | $2.5\times10^2$ | | Wang et al. (2017) | Q | 80, 240 |
| MCM:HO7CO10C12 | $3.7\times10^1$ | | Wang et al. (2017) | Q | 80, 238 |
| $C_{12}H_{24}O_2$ | $5.1\times10^2$ | | Wang et al. (2017) | Q | 80, 239 |
| AJDNADCNQXGLCC-UHFFFAOYSA-N | $4.3\times10^1$ | | Wang et al. (2017) | Q | 80, 240 |
| MCM:C131OH | $5.1\times10^4$ | | Wang et al. (2017) | Q | 80, 238 |
| $C_{13}H_{22}O_3$ | $2.3\times10^5$ | | Wang et al. (2017) | Q | 80, 239 |
| AONUZHGLRYBQFR-UHFFFAOYSA-N | $1.0\times10^5$ | | Wang et al. (2017) | Q | 80, 240 |
| MCM:C132OH | $5.1\times10^6$ | | Wang et al. (2017) | Q | 80, 238 |
| $C_{13}H_{22}O_4$ | $4.1\times10^8$ | | Wang et al. (2017) | Q | 80, 239 |
| FIXMRMMYQTVSPU-UHFFFAOYSA-N | $1.0\times10^6$ | | Wang et al. (2017) | Q | 80, 240 |
| MCM:C132OOH | $2.9\times10^9$ | | Wang et al. (2017) | Q | 80, 238 |
| $C_{13}H_{22}O_5$ | $1.7\times10^8$ | | Wang et al. (2017) | Q | 80, 239 |
| VUANIPKMYHABIK-UHFFFAOYSA-N | $1.1\times10^6$ | | Wang et al. (2017) | Q | 80, 240 |
| MCM:C133CO | $7.1\times10^9$ | | Wang et al. (2017) | Q | 80, 238 |
| $C_{13}H_{20}O_5$ | $4.6\times10^9$ | | Wang et al. (2017) | Q | 80, 239 |
| NSPNCPQPMIIWRH-UHFFFAOYSA-N | $4.7\times10^6$ | | Wang et al. (2017) | Q | 80, 240 |
| MCM:C133OH | $3.2\times10^{10}$ | | Wang et al. (2017) | Q | 80, 238 |
| $C_{13}H_{22}O_5$ | $6.0\times10^{10}$ | | Wang et al. (2017) | Q | 80, 239 |
| PJPMMZIOBGUFRQ-UHFFFAOYSA-N | $8.9\times10^4$ | | Wang et al. (2017) | Q | 80, 240 |
| MCM:C133OOH | $8.5\times10^{11}$ | | Wang et al. (2017) | Q | 80, 238 |
| $C_{13}H_{22}O_6$ | $3.7\times10^{10}$ | | Wang et al. (2017) | Q | 80, 239 |
| TWNMHUJEMNHLMV-UHFFFAOYSA-N | $5.1\times10^5$ | | Wang et al. (2017) | Q | 80, 240 |
| MCM:C134CO | $4.7\times10^{12}$ | | Wang et al. (2017) | Q | 80, 238 |
| $C_{13}H_{18}O_6$ | $2.3\times10^{11}$ | | Wang et al. (2017) | Q | 80, 239 |
| XVPKZNMZKWIEGC-UHFFFAOYSA-N | $2.1\times10^6$ | | Wang et al. (2017) | Q | 80, 240 |
| MCM:C134OH | $1.0\times10^{12}$ | | Wang et al. (2017) | Q | 80, 238 |
| $C_{13}H_{20}O_6$ | $1.1\times10^{13}$ | | Wang et al. (2017) | Q | 80, 239 |
| CKMYGJKMMJWXII-UHFFFAOYSA-N | $8.7\times10^6$ | | Wang et al. (2017) | Q | 80, 240 |
| MCM:C134OOH | $6.8\times10^{14}$ | | Wang et al. (2017) | Q | 80, 238 |
| $C_{13}H_{20}O_7$ | $1.7\times10^{12}$ | | Wang et al. (2017) | Q | 80, 239 |
| RYVAPXIZXOQMSU-UHFFFAOYSA-N | $2.3\times10^8$ | | Wang et al. (2017) | Q | 80, 240 |





Table A3.6: Ketones (RCOR) (... continued)

| Substance Formula (Trivial Name) [CAS Registry Number] InChIKey | $H_s^{cp}$ (at $T^\ominus$) $\left[\dfrac{\text{mol}}{\text{m}^3\,\text{Pa}}\right]$ | $\dfrac{\text{d}\ln H_s^{cp}}{\text{d}(1/T)}$ [K] | Reference | Type | Note |
|---|---|---|---|---|---|
| MCM:C135OOH | $5.0\times10^{17}$ | | Wang et al. (2017) | Q | 80, 238 |
| $C_{13}H_{18}O_8$ | $6.6\times10^{13}$ | | Wang et al. (2017) | Q | 80, 239 |
| VAECXYXKQRKKGE-UHFFFAOYSA-N | $9.6\times10^{6}$ | | Wang et al. (2017) | Q | 80, 240 |
| MCM:BCKAOH | $2.0\times10^{5}$ | | Wang et al. (2017) | Q | 80, 238 |
| $C_{14}H_{24}O_3$ | $8.3\times10^{6}$ | | Wang et al. (2017) | Q | 80, 239 |
| ARSAOHGCYSGONM-UHFFFAOYSA-N | $7.6\times10^{6}$ | | Wang et al. (2017) | Q | 80, 240 |
| MCM:BCKAOOH | $1.4\times10^{7}$ | | Wang et al. (2017) | Q | 80, 238 |
| $C_{14}H_{24}O_4$ | $4.0\times10^{6}$ | | Wang et al. (2017) | Q | 80, 239 |
| OXLVMLPVWDEZIL-UHFFFAOYSA-N | $7.8\times10^{7}$ | | Wang et al. (2017) | Q | 80, 240 |
| MCM:BCKBCO | $5.3\times10^{3}$ | | Wang et al. (2017) | Q | 80, 238 |
| $C_{14}H_{22}O_3$ | $7.6\times10^{5}$ | | Wang et al. (2017) | Q | 80, 239 |
| IWYBCVOMQFEOMA-UHFFFAOYSA-N | $2.7\times10^{4}$ | | Wang et al. (2017) | Q | 80, 240 |
| MCM:BCKBOOH | $1.4\times10^{7}$ | | Wang et al. (2017) | Q | 80, 238 |
| $C_{14}H_{24}O_4$ | $2.6\times10^{6}$ | | Wang et al. (2017) | Q | 80, 239 |
| SISQWITWYDFXLW-UHFFFAOYSA-N | $3.7\times10^{7}$ | | Wang et al. (2017) | Q | 80, 240 |
| MCM:C141OH | $1.4\times10^{2}$ | | Wang et al. (2017) | Q | 80, 238 |
| $C_{14}H_{24}O_2$ | $2.3\times10^{3}$ | | Wang et al. (2017) | Q | 80, 239 |
| OMNNANPOKVZFEY-UHFFFAOYSA-N | $7.8\times10^{2}$ | | Wang et al. (2017) | Q | 80, 240 |
| MCM:C142OH | $2.7\times10^{5}$ | | Wang et al. (2017) | Q | 80, 238 |
| $C_{14}H_{24}O_3$ | $8.5\times10^{6}$ | | Wang et al. (2017) | Q | 80, 239 |
| TZIWCZUVZZOIAP-UHFFFAOYSA-N | $2.0\times10^{5}$ | | Wang et al. (2017) | Q | 80, 240 |
| MCM:C142OOH | $7.4\times10^{6}$ | | Wang et al. (2017) | Q | 80, 238 |
| $C_{14}H_{24}O_4$ | $1.8\times10^{7}$ | | Wang et al. (2017) | Q | 80, 239 |
| OJVOLXXWTGQGEJ-UHFFFAOYSA-N | $1.9\times10^{6}$ | | Wang et al. (2017) | Q | 80, 240 |
| MCM:C143CO | $2.0\times10^{7}$ | | Wang et al. (2017) | Q | 80, 238 |
| $C_{14}H_{22}O_4$ | $5.1\times10^{8}$ | | Wang et al. (2017) | Q | 80, 239 |
| IPHIBGAHOYIGTO-UHFFFAOYSA-N | $5.4\times10^{6}$ | | Wang et al. (2017) | Q | 80, 240 |
| MCM:C143OH | $9.1\times10^{7}$ | | Wang et al. (2017) | Q | 80, 238 |
| $C_{14}H_{24}O_4$ | $5.0\times10^{9}$ | | Wang et al. (2017) | Q | 80, 239 |
| PUMKKTACSBJGTI-UHFFFAOYSA-N | $2.3\times10^{7}$ | | Wang et al. (2017) | Q | 80, 240 |
| MCM:C143OOH | $2.5\times10^{9}$ | | Wang et al. (2017) | Q | 80, 238 |
| $C_{14}H_{24}O_5$ | $3.4\times10^{9}$ | | Wang et al. (2017) | Q | 80, 239 |
| FMXAUTXFPHKMFA-UHFFFAOYSA-N | $1.0\times10^{6}$ | | Wang et al. (2017) | Q | 80, 240 |
| MCM:C145OH | $1.0\times10^{12}$ | | Wang et al. (2017) | Q | 80, 238 |
| $C_{14}H_{24}O_6$ | $3.1\times10^{13}$ | | Wang et al. (2017) | Q | 80, 239 |
| CADDZNHIARUDOF-UHFFFAOYSA-N | $1.5\times10^{9}$ | | Wang et al. (2017) | Q | 80, 240 |
| MCM:C145OOH | $1.5\times10^{15}$ | | Wang et al. (2017) | Q | 80, 238 |
| $C_{14}H_{24}O_7$ | $4.9\times10^{13}$ | | Wang et al. (2017) | Q | 80, 239 |
| CEBXMSFHIDLJKH-UHFFFAOYSA-N | $1.8\times10^{10}$ | | Wang et al. (2017) | Q | 80, 240 |



Table A3.6: Ketones (RCOR) (... continued)

| Substance Formula (Trivial Name) [CAS Registry Number] InChIKey | $H_s^{cp}$ (at $T^\ominus$) $\left[\dfrac{\mathrm{mol}}{\mathrm{m}^3\,\mathrm{Pa}}\right]$ | $\dfrac{\mathrm{d}\ln H_s^{cp}}{\mathrm{d}(1/T)}$ [K] | Reference | Type | Note |
|---|---|---|---|---|---|
| MCM:BCBCO | $1.4\times10^1$ | | Wang et al. (2017) | Q | 80, 238 |
| $C_{15}H_{24}O_2$ | $6.5\times10^2$ | | Wang et al. (2017) | Q | 80, 239 |
| VLNGGDKMXDHPMK-UHFFFAOYSA-N | $8.5\times10^1$ | | Wang et al. (2017) | Q | 80, 240 |
| MCM:PHCOMEOH | $3.0\times10^2$ | | Wang et al. (2017) | Q | 80, 238 |
| $C_8H_8O_2$ | $1.2\times10^3$ | | Wang et al. (2017) | Q | 80, 239 |
| ZWVHTXAYIKBMEE-UHFFFAOYSA-N | $6.0\times10^1$ | | Wang et al. (2017) | Q | 80, 240 |
| MCM:MPHCOMEOH | $1.8\times10^2$ | | Wang et al. (2017) | Q | 80, 238 |
| $C_9H_{10}O_2$ | $1.3\times10^3$ | | Wang et al. (2017) | Q | 80, 239 |
| CEJINNSYZFLSCS-UHFFFAOYSA-N | $4.2\times10^1$ | | Wang et al. (2017) | Q | 80, 240 |
| MCM:PHCOETOH | $2.8\times10^2$ | | Wang et al. (2017) | Q | 80, 238 |
| $C_9H_{10}O_2$ | $7.4\times10^2$ | | Wang et al. (2017) | Q | 80, 239 |
| WLVPRARCUSRDNI-UHFFFAOYSA-N | $6.9\times10^1$ | | Wang et al. (2017) | Q | 80, 240 |
| MCM:DMPHCOMOH | $1.0\times10^2$ | | Wang et al. (2017) | Q | 80, 238 |
| $C_{10}H_{12}O_2$ | $7.8\times10^2$ | | Wang et al. (2017) | Q | 80, 239 |
| DBYMZNOFJHQFFD-UHFFFAOYSA-N | $1.0\times10^2$ | | Wang et al. (2017) | Q | 80, 240 |
| MCM:EMPHCOMOH | $8.3\times10^1$ | | Wang et al. (2017) | Q | 80, 238 |
| $C_{11}H_{14}O_2$ | $4.5\times10^2$ | | Wang et al. (2017) | Q | 80, 239 |
| MHJXTDDYZBYNEF-UHFFFAOYSA-N | $8.5\times10^1$ | | Wang et al. (2017) | Q | 80, 240 |
| MCM:ALCOCH2OOH | $4.1\times10^6$ | | Wang et al. (2017) | Q | 80, 238 |
| $C_3H_4O_4$ | $7.6\times10^3$ | | Wang et al. (2017) | Q | 80, 239 |
| RQBGWFCHIWSUOK-UHFFFAOYSA-N | $3.0\times10^1$ | | Wang et al. (2017) | Q | 80, 240 |
| MCM:C33CO | $4.8\times10^4$ | | Wang et al. (2017) | Q | 80, 238 |
| $C_3H_2O_3$ | $2.8\times10^3$ | | Wang et al. (2017) | Q | 80, 239 |
| ICQNCHSXWNQIHC-UHFFFAOYSA-N | $4.3\times10^{-2}$ | | Wang et al. (2017) | Q | 80, 240 |
| MCM:C312COCO3H | $4.7\times10^7$ | | Wang et al. (2017) | Q | 80, 238 |
| $C_4H_4O_5$ | $3.2\times10^5$ | | Wang et al. (2017) | Q | 80, 239 |
| GUGBJTDAGNLAJO-UHFFFAOYSA-N | $8.0\times10^1$ | | Wang et al. (2017) | Q | 80, 240 |
| MCM:C413COOOH | $3.6\times10^6$ | | Wang et al. (2017) | Q | 80, 238 |
| $C_4H_6O_4$ | $3.5\times10^4$ | | Wang et al. (2017) | Q | 80, 239 |
| XUKQQLOKOLCMAI-UHFFFAOYSA-N | $4.6\times10^2$ | | Wang et al. (2017) | Q | 80, 240 |
| MCM:C4CO2OOH | $3.6\times10^6$ | | Wang et al. (2017) | Q | 80, 238 |
| $C_4H_6O_4$ | $2.6\times10^4$ | | Wang et al. (2017) | Q | 80, 239 |
| NOBUEMWCPQFCCN-UHFFFAOYSA-N | $2.7\times10^1$ | | Wang et al. (2017) | Q | 80, 240 |
| MCM:C4CODIAL | $4.3\times10^4$ | | Wang et al. (2017) | Q | 80, 238 |
| $C_4H_4O_3$ | $5.0\times10^3$ | | Wang et al. (2017) | Q | 80, 239 |
| OWIMZZMOFXSCLT-UHFFFAOYSA-N | $2.5$ | | Wang et al. (2017) | Q | 80, 240 |
| MCM:CO23C3CHO | $3.3\times10^4$ | | Wang et al. (2017) | Q | 80, 238 |
| $C_4H_4O_3$ | $3.3\times10^3$ | | Wang et al. (2017) | Q | 80, 239 |
| LVZOZHOAAHWEOQ-UHFFFAOYSA-N | $1.7\times10^{-1}$ | | Wang et al. (2017) | Q | 80, 240 |





Table A3.6: Ketones (RCOR) (...continued)

| Substance<br>Formula<br>(Trivial Name)<br>[CAS Registry Number]<br>InChIKey | $H_s^{cp}$<br>(at $T^\ominus$)<br>$\left[\dfrac{\mathrm{mol}}{\mathrm{m}^3\,\mathrm{Pa}}\right]$ | $\dfrac{\mathrm{d}\ln H_s^{cp}}{\mathrm{d}(1/T)}$<br><br>[K] | Reference | Type | Note |
|---|---|---|---|---|---|
| MCM:CO2C3CHO | $4.7\times10^1$ | | Wang et al. (2017) | Q | 80, 238 |
| $C_4H_6O_2$ | $6.0\times10^1$ | | Wang et al. (2017) | Q | 80, 239 |
| PKQIDSVLSKFZQC-UHFFFAOYSA-N | $1.7\times10^1$ | | Wang et al. (2017) | Q | 80, 240 |
| MCM:CO2C4DIAL | $2.8\times10^7$ | | Wang et al. (2017) | Q | 80, 238 |
| $C_4H_2O_4$ | $5.0\times10^5$ | | Wang et al. (2017) | Q | 80, 239 |
| OURVMEXCDVXLPR-UHFFFAOYSA-N | $1.0\times10^{-2}$ | | Wang et al. (2017) | Q | 80, 240 |
| MCM:EGLYOX | $4.7\times10^1$ | | Wang et al. (2017) | Q | 80, 238 |
| $C_4H_6O_2$ | $8.0$ | | Wang et al. (2017) | Q | 80, 239 |
| RWHQMRRVZJSKGX-UHFFFAOYSA-N | $2.3\times10^{-2}$ | | Wang et al. (2017) | Q | 80, 240 |
| MCM:VGLYOX | $1.4\times10^2$ | | Wang et al. (2017) | Q | 80, 238 |
| $C_4H_4O_2$ | $2.6\times10^1$ | | Wang et al. (2017) | Q | 80, 239 |
| SDQVYUNYDAWYIK-UHFFFAOYSA-N | $3.5\times10^{-2}$ | | Wang et al. (2017) | Q | 80, 240 |
| MCM:C4CO2DCO3H | $1.7\times10^8$ | | Wang et al. (2017) | Q | 80, 238 |
| $C_5H_4O_5$ | $5.1\times10^6$ | | Wang et al. (2017) | Q | 80, 239 |
| AXIVSVGHKYPZBE-UHFFFAOYSA-N | $8.1$ | | Wang et al. (2017) | Q | 80, 240 |
| MCM:C4MCO2OOH | $2.0\times10^6$ | | Wang et al. (2017) | Q | 80, 238 |
| $C_5H_8O_4$ | $3.6\times10^3$ | | Wang et al. (2017) | Q | 80, 239 |
| RKGLOUQDKXPRQL-UHFFFAOYSA-N | $6.3$ | | Wang et al. (2017) | Q | 80, 240 |
| MCM:C511OOH | $3.3\times10^6$ | | Wang et al. (2017) | Q | 80, 238 |
| $C_5H_8O_4$ | $7.8\times10^4$ | | Wang et al. (2017) | Q | 80, 239 |
| NBNOBTZXOPWFTP-UHFFFAOYSA-N | $1.8\times10^3$ | | Wang et al. (2017) | Q | 80, 240 |
| MCM:C5124COOOH | $2.0\times10^9$ | | Wang et al. (2017) | Q | 80, 238 |
| $C_5H_6O_5$ | $3.0\times10^6$ | | Wang et al. (2017) | Q | 80, 239 |
| BXDCDQOWFJMSCY-UHFFFAOYSA-N | $1.1\times10^2$ | | Wang et al. (2017) | Q | 80, 240 |
| MCM:C512OOH | $2.8\times10^6$ | | Wang et al. (2017) | Q | 80, 238 |
| $C_5H_8O_4$ | $9.3\times10^5$ | | Wang et al. (2017) | Q | 80, 239 |
| UGOJGQBYBBLKDX-UHFFFAOYSA-N | $2.0\times10^3$ | | Wang et al. (2017) | Q | 80, 240 |
| MCM:C515CO | $2.3\times10^7$ | | Wang et al. (2017) | Q | 80, 238 |
| $C_5H_4O_4$ | $1.0\times10^6$ | | Wang et al. (2017) | Q | 80, 239 |
| OONLUXUUWPUJOU-UHFFFAOYSA-N | $1.6\times10^1$ | | Wang et al. (2017) | Q | 80, 240 |
| MCM:C515OOH | $2.0\times10^9$ | | Wang et al. (2017) | Q | 80, 238 |
| $C_5H_6O_5$ | $3.5\times10^6$ | | Wang et al. (2017) | Q | 80, 239 |
| KCJJHAINVUWKSI-UHFFFAOYSA-N | $1.3\times10^3$ | | Wang et al. (2017) | Q | 80, 240 |
| MCM:C54CO | $2.0\times10^7$ | | Wang et al. (2017) | Q | 80, 238 |
| $C_5H_4O_4$ | $5.8\times10^5$ | | Wang et al. (2017) | Q | 80, 239 |
| KDLDWLXQVYMIBX-UHFFFAOYSA-N | $5.8\times10^{-2}$ | | Wang et al. (2017) | Q | 80, 240 |
| MCM:C5CO23CHO | $2.6\times10^4$ | | Wang et al. (2017) | Q | 80, 238 |
| $C_5H_6O_3$ | $1.4\times10^3$ | | Wang et al. (2017) | Q | 80, 239 |
| IPHREVVCPNNUEQ-UHFFFAOYSA-N | $1.1\times10^{-1}$ | | Wang et al. (2017) | Q | 80, 240 |



Table A3.6: Ketones (RCOR) (...continued)

| Substance<br>Formula<br>(Trivial Name)<br>[CAS Registry Number]<br>InChIKey | $H_s^{cp}$<br>(at $T^\ominus$)<br>$\left[\dfrac{\text{mol}}{\text{m}^3\,\text{Pa}}\right]$ | $\dfrac{\mathrm{d}\ln H_s^{cp}}{\mathrm{d}(1/T)}$<br><br>[K] | Reference | Type | Note |
|---|---|---|---|---|---|
| MCM:C5COCHOOOH | $2.8\times10^6$ | | Wang et al. (2017) | Q | 80, 238 |
| $C_5H_8O_4$ | $2.3\times10^5$ | | Wang et al. (2017) | Q | 80, 239 |
| QJMOTDLHJFXUFN-UHFFFAOYSA-N | $5.0\times10^3$ | | Wang et al. (2017) | Q | 80, 240 |
| MCM:C5DIALCO | $1.4\times10^5$ | | Wang et al. (2017) | Q | 80, 238 |
| $C_5H_4O_3$ | $4.9\times10^4$ | | Wang et al. (2017) | Q | 80, 239 |
| SFSCHQJUYKUKJM-UHFFFAOYSA-N | 1.5 | | Wang et al. (2017) | Q | 80, 240 |
| MCM:C5DICARB | $1.6\times10^2$ | | Wang et al. (2017) | Q | 80, 238 |
| $C_5H_6O_2$ | $1.1\times10^3$ | | Wang et al. (2017) | Q | 80, 239 |
| GBLMMVFQENXAFZ-UHFFFAOYSA-N | $2.7\times10^1$ | | Wang et al. (2017) | Q | 80, 240 |
| MCM:CHOC3COOOH | $3.8\times10^7$ | | Wang et al. (2017) | Q | 80, 238 |
| $C_5H_6O_5$ | $7.4\times10^5$ | | Wang et al. (2017) | Q | 80, 239 |
| KFSMXRNQWYDJRM-UHFFFAOYSA-N | $3.0\times10^2$ | | Wang et al. (2017) | Q | 80, 240 |
| MCM:CO12C4CHO | $3.6\times10^4$ | | Wang et al. (2017) | Q | 80, 238 |
| $C_5H_6O_3$ | $9.3\times10^3$ | | Wang et al. (2017) | Q | 80, 239 |
| VUSMENNBCBGZQI-UHFFFAOYSA-N | $1.5\times10^1$ | | Wang et al. (2017) | Q | 80, 240 |
| MCM:CO13C4CHO | $3.6\times10^4$ | | Wang et al. (2017) | Q | 80, 238 |
| $C_5H_6O_3$ | $6.5\times10^3$ | | Wang et al. (2017) | Q | 80, 239 |
| SSMCSEDSANTCBB-UHFFFAOYSA-N | $1.7\times10^2$ | | Wang et al. (2017) | Q | 80, 240 |
| MCM:CO23C4CHO | $2.6\times10^4$ | | Wang et al. (2017) | Q | 80, 238 |
| $C_5H_6O_3$ | $3.8\times10^3$ | | Wang et al. (2017) | Q | 80, 239 |
| QGGCMCHYJIPQCE-UHFFFAOYSA-N | 7.8 | | Wang et al. (2017) | Q | 80, 240 |
| MCM:CO24C4CHO | $2.6\times10^4$ | | Wang et al. (2017) | Q | 80, 238 |
| $C_5H_6O_3$ | $4.3\times10^3$ | | Wang et al. (2017) | Q | 80, 239 |
| XHTYNUBGBUBFOA-UHFFFAOYSA-N | 9.3 | | Wang et al. (2017) | Q | 80, 240 |
| MCM:CO2C43CHO | $4.4\times10^1$ | | Wang et al. (2017) | Q | 80, 238 |
| $C_5H_8O_2$ | $1.7\times10^1$ | | Wang et al. (2017) | Q | 80, 239 |
| KNTLTMLEQPLVDA-UHFFFAOYSA-N | 4.9 | | Wang et al. (2017) | Q | 80, 240 |
| MCM:CO2C4CHO | $5.5\times10^2$ | 8400 | Wieser et al. (2023) | Q | 437 |
| $C_5H_8O_2$ | $3.9\times10^1$ | | Wang et al. (2017) | Q | 80, 238 |
| KEHNRUNQZGRQHU-UHFFFAOYSA-N | $3.6\times10^2$ | | Wang et al. (2017) | Q | 80, 239 |
| | $7.6\times10^1$ | | Wang et al. (2017) | Q | 80, 240 |
| MCM:CO3C4CHO | $3.9\times10^1$ | | Wang et al. (2017) | Q | 80, 238 |
| $C_5H_8O_2$ | $2.6\times10^1$ | | Wang et al. (2017) | Q | 80, 239 |
| ZNNXJRURXWWGLN-UHFFFAOYSA-N | 6.5 | | Wang et al. (2017) | Q | 80, 240 |
| MCM:IPRGLYOX | $4.4\times10^1$ | | Wang et al. (2017) | Q | 80, 238 |
| $C_5H_8O_2$ | 4.9 | | Wang et al. (2017) | Q | 80, 239 |
| FTDZDHIBENIBKZ-UHFFFAOYSA-N | $1.9\times10^{-2}$ | | Wang et al. (2017) | Q | 80, 240 |
| MCM:PGLYOX | $3.9\times10^1$ | | Wang et al. (2017) | Q | 80, 238 |
| $C_5H_8O_2$ | 5.8 | | Wang et al. (2017) | Q | 80, 239 |
| GDTHVMAIBQVUMV-UHFFFAOYSA-N | $1.9\times10^{-2}$ | | Wang et al. (2017) | Q | 80, 240 |





Table A3.6: Ketones (RCOR) (...continued)

| Substance<br>Formula<br>(Trivial Name)<br>[CAS Registry Number]<br>InChIKey | $H_s^{cp}$<br>(at $T^{\ominus}$)<br>$\left[\dfrac{\text{mol}}{\text{m}^3\,\text{Pa}}\right]$ | $\dfrac{\text{d}\ln H_s^{cp}}{\text{d}(1/T)}$<br><br>[K] | Reference | Type | Note |
|---|---|---|---|---|---|
| MCM:C23C54CHO<br>$C_6H_8O_3$<br>CBIOWOAHGWGTSI-UHFFFAOYSA-N | $2.7\times10^4$<br>$7.4\times10^2$<br>2.8 | | Wang et al. (2017)<br>Wang et al. (2017)<br>Wang et al. (2017) | Q<br>Q<br>Q | 80, 238<br>80, 239<br>80, 240 |
| MCM:C3COCCHO<br>$C_6H_{10}O_2$<br>PVLKSJIUOKAUMV-UHFFFAOYSA-N | $3.0\times10^1$<br>$2.2\times10^1$<br>7.6 | | Wang et al. (2017)<br>Wang et al. (2017)<br>Wang et al. (2017) | Q<br>Q<br>Q | 80, 238<br>80, 239<br>80, 240 |
| MCM:C45IC5CHO<br>$C_6H_8O_3$<br>WETHXHAOPNKQLZ-UHFFFAOYSA-N | $2.7\times10^4$<br>$8.0\times10^2$<br>$4.5\times10^{-2}$ | | Wang et al. (2017)<br>Wang et al. (2017)<br>Wang et al. (2017) | Q<br>Q<br>Q | 80, 238<br>80, 239<br>80, 240 |
| MCM:C4ECO2OOH<br>$C_6H_{10}O_4$<br>CMRDMVGIWXQQFX-UHFFFAOYSA-N | $1.8\times10^6$<br>$1.8\times10^3$<br>3.6 | | Wang et al. (2017)<br>Wang et al. (2017)<br>Wang et al. (2017) | Q<br>Q<br>Q | 80, 238<br>80, 239<br>80, 240 |
| MCM:C511CHO<br>$C_6H_8O_3$<br>SLKPDJWOVVFRFW-UHFFFAOYSA-N | $3.2\times10^4$<br>$1.4\times10^4$<br>$1.8\times10^2$ | | Wang et al. (2017)<br>Wang et al. (2017)<br>Wang et al. (2017) | Q<br>Q<br>Q | 80, 238<br>80, 239<br>80, 240 |
| MCM:C511CO3H<br>$C_6H_8O_5$<br>PKVJBWZRBQBAMT-UHFFFAOYSA-N | $3.5\times10^7$<br>$1.6\times10^6$<br>$1.5\times10^2$ | | Wang et al. (2017)<br>Wang et al. (2017)<br>Wang et al. (2017) | Q<br>Q<br>Q | 80, 238<br>80, 239<br>80, 240 |
| MCM:C512CO3H<br>$C_6H_8O_5$<br>OPLKNVAQLNTMAR-UHFFFAOYSA-N | $3.4\times10^7$<br>$1.6\times10^6$<br>$2.0\times10^3$ | | Wang et al. (2017)<br>Wang et al. (2017)<br>Wang et al. (2017) | Q<br>Q<br>Q | 80, 238<br>80, 239<br>80, 240 |
| MCM:C515CHO<br>$C_6H_6O_4$<br>JHVNJBQABGTRPH-UHFFFAOYSA-N | $2.0\times10^7$<br>$8.7\times10^5$<br>$1.6\times10^2$ | | Wang et al. (2017)<br>Wang et al. (2017)<br>Wang et al. (2017) | Q<br>Q<br>Q | 80, 238<br>80, 239<br>80, 240 |
| MCM:C515CO3H<br>$C_6H_6O_6$<br>XPLLKUAMAYNXAT-UHFFFAOYSA-N | $2.2\times10^{10}$<br>$1.1\times10^8$<br>$2.3\times10^3$ | | Wang et al. (2017)<br>Wang et al. (2017)<br>Wang et al. (2017) | Q<br>Q<br>Q | 80, 238<br>80, 239<br>80, 240 |
| MCM:C5CO2DCO3H<br>$C_6H_6O_5$<br>DHHXAZIZMIZDQV-UHFFFAOYSA-N | $1.1\times10^8$<br>$4.5\times10^6$<br>$3.7\times10^{-1}$ | | Wang et al. (2017)<br>Wang et al. (2017)<br>Wang et al. (2017) | Q<br>Q<br>Q | 80, 238<br>80, 239<br>80, 240 |
| MCM:C5DCO2CO3H<br>$C_6H_6O_5$<br>GJOWGOOSLUXRQO-UHFFFAOYSA-N | $1.1\times10^8$<br>$4.2\times10^6$<br>$6.8\times10^{-1}$ | | Wang et al. (2017)<br>Wang et al. (2017)<br>Wang et al. (2017) | Q<br>Q<br>Q | 80, 238<br>80, 239<br>80, 240 |
| MCM:C5MDICARB<br>$C_6H_8O_2$<br>PFLPZDBOPDYOJU-UHFFFAOYSA-N | $1.0\times10^2$<br>$1.0\times10^3$<br>5.6 | | Wang et al. (2017)<br>Wang et al. (2017)<br>Wang et al. (2017) | Q<br>Q<br>Q | 80, 238<br>80, 239<br>80, 240 |
| MCM:C5TRONCO3H<br>$C_6H_6O_6$<br>GGHLQXKHXYQGCE-UHFFFAOYSA-N | $2.2\times10^{10}$<br>$4.6\times10^7$<br>$1.9\times10^2$ | | Wang et al. (2017)<br>Wang et al. (2017)<br>Wang et al. (2017) | Q<br>Q<br>Q | 80, 238<br>80, 239<br>80, 240 |




Table A3.6: Ketones (RCOR) (...continued)

| Substance Formula (Trivial Name) [CAS Registry Number] InChIKey | $H_s^{cp}$ (at $T^{\ominus}$) $\left[\dfrac{\mathrm{mol}}{\mathrm{m^3\,Pa}}\right]$ | $\dfrac{\mathrm{d}\ln H_s^{cp}}{\mathrm{d}(1/T)}$ [K] | Reference | Type | Note |
|---|---|---|---|---|---|
| MCM:C6125CO | $9.3\times10^4$ | | Wang et al. (2017) | Q | 80, 238 |
| $C_6H_6O_3$ | $6.2\times10^4$ | | Wang et al. (2017) | Q | 80, 239 |
| RFTMILUWMDIPHH-UHFFFAOYSA-N | 5.4 | | Wang et al. (2017) | Q | 80, 240 |
| MCM:C6135COOOH | $1.7\times10^9$ | | Wang et al. (2017) | Q | 80, 238 |
| $C_6H_8O_5$ | $4.8\times10^6$ | | Wang et al. (2017) | Q | 80, 239 |
| HGYTXVUJMUREIP-UHFFFAOYSA-N | $2.0\times10^4$ | | Wang et al. (2017) | Q | 80, 240 |
| MCM:C6145COOOH | $1.7\times10^9$ | | Wang et al. (2017) | Q | 80, 238 |
| $C_6H_8O_5$ | $7.8\times10^6$ | | Wang et al. (2017) | Q | 80, 239 |
| DSJLQKNTJKMXRQ-UHFFFAOYSA-N | $2.6\times10^3$ | | Wang et al. (2017) | Q | 80, 240 |
| MCM:C615CO2OOH | $1.1\times10^7$ | | Wang et al. (2017) | Q | 80, 238 |
| $C_6H_8O_4$ | $1.2\times10^5$ | | Wang et al. (2017) | Q | 80, 239 |
| MOFLPBJDWVVKPL-UHFFFAOYSA-N | $2.3\times10^4$ | | Wang et al. (2017) | Q | 80, 240 |
| MCM:C615CO | $2.2\times10^4$ | | Wang et al. (2017) | Q | 80, 238 |
| $C_6H_8O_3$ | $2.8\times10^2$ | | Wang et al. (2017) | Q | 80, 239 |
| BRRWJVYIMRBLKB-UHFFFAOYSA-N | 1.2 | | Wang et al. (2017) | Q | 80, 240 |
| MCM:C616OOH | $2.4\times10^9$ | | Wang et al. (2017) | Q | 80, 238 |
| $C_6H_8O_5$ | $1.7\times10^7$ | | Wang et al. (2017) | Q | 80, 239 |
| RZNSBZRPHHAABM-UHFFFAOYSA-N | $3.8\times10^3$ | | Wang et al. (2017) | Q | 80, 240 |
| MCM:C617OOH | $1.8\times10^6$ | | Wang et al. (2017) | Q | 80, 238 |
| $C_6H_{10}O_4$ | $6.2\times10^3$ | | Wang et al. (2017) | Q | 80, 239 |
| VPKXUGZDYIOAPR-UHFFFAOYSA-N | $4.8\times10^1$ | | Wang et al. (2017) | Q | 80, 240 |
| MCM:C618OOH | $1.8\times10^6$ | | Wang et al. (2017) | Q | 80, 238 |
| $C_6H_{10}O_4$ | $2.8\times10^3$ | | Wang et al. (2017) | Q | 80, 239 |
| AMQFKFSLRYYJMS-UHFFFAOYSA-N | $1.6\times10^2$ | | Wang et al. (2017) | Q | 80, 240 |
| MCM:C620OOH | $2.4\times10^9$ | | Wang et al. (2017) | Q | 80, 238 |
| $C_6H_8O_5$ | $1.0\times10^7$ | | Wang et al. (2017) | Q | 80, 239 |
| UWWRYJPKFSIPFS-UHFFFAOYSA-N | $1.4\times10^3$ | | Wang et al. (2017) | Q | 80, 240 |
| MCM:C626OOH | $2.6\times10^6$ | | Wang et al. (2017) | Q | 80, 238 |
| $C_6H_{10}O_4$ | $2.3\times10^6$ | | Wang et al. (2017) | Q | 80, 239 |
| DHTMLHKAZOWLFT-UHFFFAOYSA-N | $4.6\times10^3$ | | Wang et al. (2017) | Q | 80, 240 |
| MCM:C65OOH | $1.8\times10^6$ | | Wang et al. (2017) | Q | 80, 238 |
| $C_6H_{10}O_4$ | $1.6\times10^4$ | | Wang et al. (2017) | Q | 80, 239 |
| LHKMAYQHAYSMQO-UHFFFAOYSA-N | $5.4\times10^2$ | | Wang et al. (2017) | Q | 80, 240 |
| MCM:C6CO134 | $2.3\times10^4$ | | Wang et al. (2017) | Q | 80, 238 |
| $C_6H_8O_3$ | $1.3\times10^3$ | | Wang et al. (2017) | Q | 80, 239 |
| VIOZHXSHLWOADQ-UHFFFAOYSA-N | 6.2 | | Wang et al. (2017) | Q | 80, 240 |
| MCM:C6CO4DB | $8.3\times10^7$ | | Wang et al. (2017) | Q | 80, 238 |
| $C_6H_4O_4$ | $2.3\times10^6$ | | Wang et al. (2017) | Q | 80, 239 |
| OGFBTMOHTIWBJU-UHFFFAOYSA-N | $1.6\times10^{-1}$ | | Wang et al. (2017) | Q | 80, 240 |





Table A3.6: Ketones (RCOR) (...continued)

| Substance<br>Formula<br>(Trivial Name)<br>[CAS Registry Number]<br>InChIKey | $H_s^{cp}$<br>(at $T^{\ominus}$)<br>$\left[\dfrac{\text{mol}}{\text{m}^3\,\text{Pa}}\right]$ | $\dfrac{\text{d}\ln H_s^{cp}}{\text{d}(1/T)}$<br><br>[K] | Reference | Type | Note |
|---|---|---|---|---|---|
| MCM:C6COALCO3H | $3.4\times10^7$ | | Wang et al. (2017) | Q | 80, 238 |
| $C_6H_8O_5$ | $4.3\times10^6$ | | Wang et al. (2017) | Q | 80, 239 |
| WRLXIVUJPLXKIG-UHFFFAOYSA-N | $6.5\times10^3$ | | Wang et al. (2017) | Q | 80, 240 |
| MCM:C6COCHOOOH | $2.3\times10^6$ | | Wang et al. (2017) | Q | 80, 238 |
| $C_6H_{10}O_4$ | $1.5\times10^6$ | | Wang et al. (2017) | Q | 80, 239 |
| DQAWORJSQGOPLV-UHFFFAOYSA-N | $5.8\times10^3$ | | Wang et al. (2017) | Q | 80, 240 |
| MCM:C6CODIAL | $2.8\times10^4$ | | Wang et al. (2017) | Q | 80, 238 |
| $C_6H_8O_3$ | $3.2\times10^4$ | | Wang et al. (2017) | Q | 80, 239 |
| PAXFMQWEAJBIJB-UHFFFAOYSA-N | $2.5\times10^3$ | | Wang et al. (2017) | Q | 80, 240 |
| MCM:C6DICARB | $1.4\times10^2$ | | Wang et al. (2017) | Q | 80, 238 |
| $C_6H_8O_2$ | $6.5\times10^2$ | | Wang et al. (2017) | Q | 80, 239 |
| GVKYFODEMNCLGS-UHFFFAOYSA-N | 5.9 | | Wang et al. (2017) | Q | 80, 240 |
| MCM:CO123C5CHO | $2.0\times10^7$ | | Wang et al. (2017) | Q | 80, 238 |
| $C_6H_6O_4$ | $1.7\times10^6$ | | Wang et al. (2017) | Q | 80, 239 |
| GLSSGXNNMIEKBR-UHFFFAOYSA-N | $1.8\times10^2$ | | Wang et al. (2017) | Q | 80, 240 |
| MCM:CO235C5CHO | $1.6\times10^7$ | | Wang et al. (2017) | Q | 80, 238 |
| $C_6H_6O_4$ | $2.1\times10^5$ | | Wang et al. (2017) | Q | 80, 239 |
| GPIFCHFPYCMSRY-UHFFFAOYSA-N | 4.7 | | Wang et al. (2017) | Q | 80, 240 |
| MCM:CO24M3CHO | $2.7\times10^4$ | | Wang et al. (2017) | Q | 80, 238 |
| $C_6H_8O_3$ | $8.0\times10^2$ | | Wang et al. (2017) | Q | 80, 239 |
| OQYGUGGUHGKPLC-UHFFFAOYSA-N | 2.7 | | Wang et al. (2017) | Q | 80, 240 |
| MCM:CO2C4GLYOX | $2.3\times10^4$ | | Wang et al. (2017) | Q | 80, 238 |
| $C_6H_8O_3$ | $7.4\times10^3$ | | Wang et al. (2017) | Q | 80, 239 |
| YEJDUVJLIVLCFZ-UHFFFAOYSA-N | $8.1\times10^1$ | | Wang et al. (2017) | Q | 80, 240 |
| MCM:CO2C54CHO | $3.6\times10^1$ | | Wang et al. (2017) | Q | 80, 238 |
| $C_6H_{10}O_2$ | $1.8\times10^2$ | | Wang et al. (2017) | Q | 80, 239 |
| IDAHIBMEKOEBRG-UHFFFAOYSA-N | $5.9\times10^1$ | | Wang et al. (2017) | Q | 80, 240 |
| MCM:CO2M33CHO | $2.5\times10^1$ | | Wang et al. (2017) | Q | 80, 238 |
| $C_6H_{10}O_2$ | $1.1\times10^1$ | | Wang et al. (2017) | Q | 80, 239 |
| HOOWKSPRVCRTJK-UHFFFAOYSA-N | 1.5 | | Wang et al. (2017) | Q | 80, 240 |
| MCM:CO2M3C4CHO | $3.6\times10^1$ | | Wang et al. (2017) | Q | 80, 238 |
| $C_6H_{10}O_2$ | $1.8\times10^2$ | | Wang et al. (2017) | Q | 80, 239 |
| GQHWESFTVGVFIA-UHFFFAOYSA-N | $4.1\times10^1$ | | Wang et al. (2017) | Q | 80, 240 |
| MCM:CO35C5CHO | $2.3\times10^4$ | | Wang et al. (2017) | Q | 80, 238 |
| $C_6H_8O_3$ | $1.4\times10^3$ | | Wang et al. (2017) | Q | 80, 239 |
| KXLMWINKDVAJKB-UHFFFAOYSA-N | 2.6 | | Wang et al. (2017) | Q | 80, 240 |
| MCM:CO3C54CHO | $3.6\times10^1$ | | Wang et al. (2017) | Q | 80, 238 |
| $C_6H_{10}O_2$ | $1.6\times10^1$ | | Wang et al. (2017) | Q | 80, 239 |
| HFHZCESKBNQESK-UHFFFAOYSA-N | 1.8 | | Wang et al. (2017) | Q | 80, 240 |





Table A3.6: Ketones (RCOR) (...continued)

| Substance<br>Formula<br>(Trivial Name)<br>[CAS Registry Number]<br>InChIKey | $H_s^{cp}$<br>(at $T^\ominus$)<br>$\left[\dfrac{\text{mol}}{\text{m}^3\,\text{Pa}}\right]$ | $\dfrac{\text{d}\ln H_s^{cp}}{\text{d}(1/T)}$<br><br>[K] | Reference | Type | Note |
|---|---|---|---|---|---|
| MCM:CO3C5CHO | $3.0\times10^1$ | | Wang et al. (2017) | Q | 80, 238 |
| $C_6H_{10}O_2$ | $2.3\times10^2$ | | Wang et al. (2017) | Q | 80, 239 |
| KKOFYQBBUSZDKJ-UHFFFAOYSA-N | $2.6\times10^1$ | | Wang et al. (2017) | Q | 80, 240 |
| MCM:CO45C5CHO | $2.3\times10^4$ | | Wang et al. (2017) | Q | 80, 238 |
| $C_6H_8O_3$ | $9.1\times10^2$ | | Wang et al. (2017) | Q | 80, 239 |
| WXTRRXXKCNKUIA-UHFFFAOYSA-N | $8.1\times10^{-2}$ | | Wang et al. (2017) | Q | 80, 240 |
| MCM:ECO4 | $1.6\times10^7$ | | Wang et al. (2017) | Q | 80, 238 |
| $C_6H_6O_4$ | $2.5\times10^5$ | | Wang et al. (2017) | Q | 80, 239 |
| RGMUISDTMIPFQK-UHFFFAOYSA-N | $4.2\times10^{-2}$ | | Wang et al. (2017) | Q | 80, 240 |
| MCM:MC5DICARB | $1.0\times10^2$ | | Wang et al. (2017) | Q | 80, 238 |
| $C_6H_8O_2$ | $8.9\times10^2$ | | Wang et al. (2017) | Q | 80, 239 |
| PEBMIJGMJUFNQE-UHFFFAOYSA-N | $2.8\times10^1$ | | Wang et al. (2017) | Q | 80, 240 |
| MCM:C4DBM2CO3H | $7.8\times10^7$ | | Wang et al. (2017) | Q | 80, 238 |
| $C_7H_8O_5$ | $5.1\times10^6$ | | Wang et al. (2017) | Q | 80, 239 |
| OAFSDIWBIPMWQP-UHFFFAOYSA-N | $8.0\times10^{-1}$ | | Wang et al. (2017) | Q | 80, 240 |
| MCM:C617CHO | $1.8\times10^4$ | | Wang et al. (2017) | Q | 80, 238 |
| $C_7H_{10}O_3$ | $4.6\times10^2$ | | Wang et al. (2017) | Q | 80, 239 |
| NFXFJRVNKKYMOX-UHFFFAOYSA-N | $7.1\times10^1$ | | Wang et al. (2017) | Q | 80, 240 |
| MCM:C617CO3H | $2.0\times10^7$ | | Wang et al. (2017) | Q | 80, 238 |
| $C_7H_{10}O_5$ | $1.2\times10^5$ | | Wang et al. (2017) | Q | 80, 239 |
| GEJLCFNTSCJUHB-UHFFFAOYSA-N | $5.6\times10^2$ | | Wang et al. (2017) | Q | 80, 240 |
| MCM:C618CO3H | $2.0\times10^7$ | | Wang et al. (2017) | Q | 80, 238 |
| $C_7H_{10}O_5$ | $1.0\times10^5$ | | Wang et al. (2017) | Q | 80, 239 |
| BCVYZYXNMFZMHQ-UHFFFAOYSA-N | $1.4\times10^3$ | | Wang et al. (2017) | Q | 80, 240 |
| MCM:C626CHO | $2.6\times10^4$ | | Wang et al. (2017) | Q | 80, 238 |
| $C_7H_{10}O_3$ | $5.4\times10^4$ | | Wang et al. (2017) | Q | 80, 239 |
| HBIBDTOPQGUCOQ-UHFFFAOYSA-N | $2.5\times10^3$ | | Wang et al. (2017) | Q | 80, 240 |
| MCM:C626CO3H | $3.2\times10^7$ | | Wang et al. (2017) | Q | 80, 238 |
| $C_7H_{10}O_5$ | $2.9\times10^6$ | | Wang et al. (2017) | Q | 80, 239 |
| XXGYBOYKTQFTKB-UHFFFAOYSA-N | $8.0\times10^3$ | | Wang et al. (2017) | Q | 80, 240 |
| MCM:C6M5CO2OOH | $6.0\times10^6$ | | Wang et al. (2017) | Q | 80, 238 |
| $C_7H_{10}O_4$ | $1.4\times10^4$ | | Wang et al. (2017) | Q | 80, 239 |
| BPEPZMFTCHTDHQ-UHFFFAOYSA-N | $1.5\times10^4$ | | Wang et al. (2017) | Q | 80, 240 |
| MCM:C715CO2OOH | $1.0\times10^7$ | | Wang et al. (2017) | Q | 80, 238 |
| $C_7H_{10}O_4$ | $5.8\times10^4$ | | Wang et al. (2017) | Q | 80, 239 |
| MXRQCKDWMJYUGS-UHFFFAOYSA-N | $1.0\times10^4$ | | Wang et al. (2017) | Q | 80, 240 |
| MCM:C716OOH | $1.6\times10^9$ | | Wang et al. (2017) | Q | 80, 238 |
| $C_7H_{10}O_5$ | $8.3\times10^6$ | | Wang et al. (2017) | Q | 80, 239 |
| PDOSKCUHLZEIAA-UHFFFAOYSA-N | $2.5\times10^4$ | | Wang et al. (2017) | Q | 80, 240 |





Table A3.6: Ketones (RCOR) (... continued)

| Substance Formula (Trivial Name) [CAS Registry Number] InChIKey | $H_s^{cp}$ (at $T^{\ominus}$) $\left[\dfrac{\text{mol}}{\text{m}^3\,\text{Pa}}\right]$ | $\dfrac{\text{d}\ln H_s^{cp}}{\text{d}(1/T)}$ [K] | Reference | Type | Note |
|---|---|---|---|---|---|
| MCM:C717OOH | $1.6\times10^9$ | | Wang et al. (2017) | Q | 80, 238 |
| $C_7H_{10}O_5$ | $4.9\times10^7$ | | Wang et al. (2017) | Q | 80, 239 |
| PIJZITIWWOWVNF-UHFFFAOYSA-N | $6.8\times10^4$ | | Wang et al. (2017) | Q | 80, 240 |
| MCM:C718OOH | $1.5\times10^6$ | | Wang et al. (2017) | Q | 80, 238 |
| $C_7H_{12}O_4$ | $2.0\times10^5$ | | Wang et al. (2017) | Q | 80, 239 |
| IUOFUPZPJIEECL-UHFFFAOYSA-N | $1.1\times10^3$ | | Wang et al. (2017) | Q | 80, 240 |
| MCM:C731OOH | $2.1\times10^6$ | | Wang et al. (2017) | Q | 80, 238 |
| $C_7H_{12}O_4$ | $4.3\times10^6$ | | Wang et al. (2017) | Q | 80, 239 |
| GGMJSZWHPXYNJN-UHFFFAOYSA-N | $4.8\times10^3$ | | Wang et al. (2017) | Q | 80, 240 |
| MCM:C735OOH | $1.4\times10^9$ | | Wang et al. (2017) | Q | 80, 238 |
| $C_7H_{10}O_5$ | $8.0\times10^6$ | | Wang et al. (2017) | Q | 80, 239 |
| FKXMDVIITUJJPH-UHFFFAOYSA-N | $1.4\times10^4$ | | Wang et al. (2017) | Q | 80, 240 |
| MCM:C736CO | $5.3\times10^4$ | | Wang et al. (2017) | Q | 80, 238 |
| $C_7H_8O_3$ | $1.4\times10^4$ | | Wang et al. (2017) | Q | 80, 239 |
| XIQVOMVDKSOEBZ-UHFFFAOYSA-N | $2.8\times10^1$ | | Wang et al. (2017) | Q | 80, 240 |
| MCM:C736OOH | $4.3\times10^6$ | | Wang et al. (2017) | Q | 80, 238 |
| $C_7H_{10}O_4$ | $1.4\times10^5$ | | Wang et al. (2017) | Q | 80, 239 |
| FLZNSMFAMOSJOT-UHFFFAOYSA-N | $4.6\times10^3$ | | Wang et al. (2017) | Q | 80, 240 |
| MCM:C7ADICARB | $6.9\times10^1$ | | Wang et al. (2017) | Q | 80, 238 |
| $C_7H_{10}O_2$ | $1.3\times10^3$ | | Wang et al. (2017) | Q | 80, 239 |
| JOIPXRIQHWMPFU-UHFFFAOYSA-N | $8.5\times10^1$ | | Wang et al. (2017) | Q | 80, 240 |
| MCM:C7CO2DBAL | $7.6\times10^4$ | | Wang et al. (2017) | Q | 80, 238 |
| $C_7H_8O_3$ | $5.5\times10^4$ | | Wang et al. (2017) | Q | 80, 239 |
| YGMUVGSOGISXBZ-UHFFFAOYSA-N | $3.1$ | | Wang et al. (2017) | Q | 80, 240 |
| MCM:C7CO2DCO3H | $9.3\times10^7$ | | Wang et al. (2017) | Q | 80, 238 |
| $C_7H_8O_5$ | $1.8\times10^6$ | | Wang et al. (2017) | Q | 80, 239 |
| GSEZJGCNNFTSDI-UHFFFAOYSA-N | $2.6\times10^{-1}$ | | Wang et al. (2017) | Q | 80, 240 |
| MCM:C7CO4DB | $5.8\times10^7$ | | Wang et al. (2017) | Q | 80, 238 |
| $C_7H_6O_4$ | $1.5\times10^6$ | | Wang et al. (2017) | Q | 80, 239 |
| RYPSCZRULGRFHH-UHFFFAOYSA-N | $4.9\times10^{-1}$ | | Wang et al. (2017) | Q | 80, 240 |
| MCM:C7DDICARB | $9.3\times10^1$ | | Wang et al. (2017) | Q | 80, 238 |
| $C_7H_{10}O_2$ | $6.2\times10^2$ | | Wang et al. (2017) | Q | 80, 239 |
| SRVPSCZJUZZLCY-UHFFFAOYSA-N | $1.1$ | | Wang et al. (2017) | Q | 80, 240 |
| MCM:C7DICARB | $1.1\times10^2$ | | Wang et al. (2017) | Q | 80, 238 |
| $C_7H_{10}O_2$ | $3.7\times10^2$ | | Wang et al. (2017) | Q | 80, 239 |
| GLRVVKVYRCSDLF-UHFFFAOYSA-N | $1.6\times10^1$ | | Wang et al. (2017) | Q | 80, 240 |
| MCM:CO235C6CHO | $1.2\times10^7$ | | Wang et al. (2017) | Q | 80, 238 |
| $C_7H_8O_4$ | $2.6\times10^5$ | | Wang et al. (2017) | Q | 80, 239 |
| DKBJFPXQZAZJBK-UHFFFAOYSA-N | $4.2\times10^2$ | | Wang et al. (2017) | Q | 80, 240 |





Table A3.6: Ketones (RCOR) (... continued)

| Substance Formula (Trivial Name) [CAS Registry Number] InChIKey | $H_s^{cp}$ (at $T^{\ominus}$) $\left[\dfrac{\mathrm{mol}}{\mathrm{m^3\,Pa}}\right]$ | $\dfrac{\mathrm{d}\ln H_s^{cp}}{\mathrm{d}(1/T)}$ [K] | Reference | Type | Note |
|---|---|---|---|---|---|
| MCM:CO25C6CHO | $1.9\times10^4$ | | Wang et al. (2017) | Q | 80, 238 |
| $C_7H_{10}O_3$ | $2.6\times10^4$ | | Wang et al. (2017) | Q | 80, 239 |
| ZTMYUKBUAYLBJN-UHFFFAOYSA-N | $4.9\times10^3$ | | Wang et al. (2017) | Q | 80, 240 |
| MCM:IC7DICARB | $1.3\times10^2$ | | Wang et al. (2017) | Q | 80, 238 |
| $C_7H_{10}O_2$ | $3.1\times10^2$ | | Wang et al. (2017) | Q | 80, 239 |
| XQQMMKHYZYQABV-UHFFFAOYSA-N | $1.1\times10^1$ | | Wang et al. (2017) | Q | 80, 240 |
| MCM:MC6CO2OOH | $6.0\times10^6$ | | Wang et al. (2017) | Q | 80, 238 |
| $C_7H_{10}O_4$ | $6.2\times10^3$ | | Wang et al. (2017) | Q | 80, 239 |
| HBFXSTFRJMUUSF-UHFFFAOYSA-N | $1.1\times10^4$ | | Wang et al. (2017) | Q | 80, 240 |
| MCM:C4DBMECO3H | $6.0\times10^7$ | | Wang et al. (2017) | Q | 80, 238 |
| $C_8H_{10}O_5$ | $2.2\times10^6$ | | Wang et al. (2017) | Q | 80, 239 |
| MRKOFGXLZVJMLJ-UHFFFAOYSA-N | $6.5\times10^{-1}$ | | Wang et al. (2017) | Q | 80, 240 |
| MCM:C6CO4M2DB | $3.9\times10^7$ | | Wang et al. (2017) | Q | 80, 238 |
| $C_8H_8O_4$ | $1.7\times10^6$ | | Wang et al. (2017) | Q | 80, 239 |
| XSTUBABFSAZOJC-UHFFFAOYSA-N | $1.7$ | | Wang et al. (2017) | Q | 80, 240 |
| MCM:C6ETCO4DB | $4.5\times10^7$ | | Wang et al. (2017) | Q | 80, 238 |
| $C_8H_8O_4$ | $8.1\times10^5$ | | Wang et al. (2017) | Q | 80, 239 |
| OAQBTICOMUFUNE-UHFFFAOYSA-N | $2.8\times10^{-1}$ | | Wang et al. (2017) | Q | 80, 240 |
| MCM:C718CO3H | $1.7\times10^7$ | | Wang et al. (2017) | Q | 80, 238 |
| $C_8H_{12}O_5$ | $2.4\times10^5$ | | Wang et al. (2017) | Q | 80, 239 |
| TVOCFMMHLDKVAZ-UHFFFAOYSA-N | $8.0\times10^2$ | | Wang et al. (2017) | Q | 80, 240 |
| MCM:C731CO3H | $2.5\times10^7$ | | Wang et al. (2017) | Q | 80, 238 |
| $C_8H_{12}O_5$ | $2.1\times10^6$ | | Wang et al. (2017) | Q | 80, 239 |
| NTZMYPVPFVPPPF-UHFFFAOYSA-N | $8.1\times10^3$ | | Wang et al. (2017) | Q | 80, 240 |
| MCM:C7M6CO2OOH | $5.4\times10^6$ | | Wang et al. (2017) | Q | 80, 238 |
| $C_8H_{12}O_4$ | $6.6\times10^3$ | | Wang et al. (2017) | Q | 80, 239 |
| SYXNCLOQMRSNCQ-UHFFFAOYSA-N | $5.5\times10^3$ | | Wang et al. (2017) | Q | 80, 240 |
| MCM:C7MCO2DBAL | $7.1\times10^4$ | | Wang et al. (2017) | Q | 80, 238 |
| $C_8H_{10}O_3$ | $3.2\times10^4$ | | Wang et al. (2017) | Q | 80, 239 |
| PDQBXAJOXSGWHJ-UHFFFAOYSA-N | $1.8$ | | Wang et al. (2017) | Q | 80, 240 |
| MCM:C7MCO2OOH | $8.9\times10^6$ | | Wang et al. (2017) | Q | 80, 238 |
| $C_8H_{12}O_4$ | $3.6\times10^4$ | | Wang et al. (2017) | Q | 80, 239 |
| IVJZKSXAPVTQOQ-UHFFFAOYSA-N | $7.1\times10^3$ | | Wang et al. (2017) | Q | 80, 240 |
| MCM:C815CO2OOH | $7.8\times10^6$ | | Wang et al. (2017) | Q | 80, 238 |
| $C_8H_{12}O_4$ | $3.2\times10^4$ | | Wang et al. (2017) | Q | 80, 239 |
| BDUHFTDKOJODDX-UHFFFAOYSA-N | $6.2\times10^3$ | | Wang et al. (2017) | Q | 80, 240 |
| MCM:C817CO | $3.1\times10^5$ | 12000 | Wieser et al. (2023) | Q | 437 |
| $C_8H_{12}O_3$ | $1.7\times10^4$ | | Wang et al. (2017) | Q | 80, 238 |
| WSCYQCAMZDXSLH-UHFFFAOYSA-N | $5.8\times10^3$ | | Wang et al. (2017) | Q | 80, 239 |
| | $3.1\times10^3$ | | Wang et al. (2017) | Q | 80, 240 |





Table A3.6: Ketones (RCOR) (...continued)

| Substance<br>Formula<br>(Trivial Name)<br>[CAS Registry Number]<br>InChIKey | $H_s^{cp}$<br>(at $T^{\ominus}$)<br>$\left[\dfrac{\text{mol}}{\text{m}^3\,\text{Pa}}\right]$ | $\dfrac{\text{d}\ln H_s^{cp}}{\text{d}(1/T)}$<br><br>[K] | Reference | Type | Note |
|---|---|---|---|---|---|
| MCM:C830CO<br>$C_8H_{12}O_2$<br>MHDCXCKIQVLVLU-UHFFFAOYSA-N | $5.9\times10^1$<br>$3.8\times10^2$<br>$1.5\times10^2$ | | Wang et al. (2017)<br>Wang et al. (2017)<br>Wang et al. (2017) | Q<br>Q<br>Q | 80, 238<br>80, 239<br>80, 240 |
| MCM:C831CO<br>$C_8H_{12}O_3$<br>ZUSGUVUFJOBUST-UHFFFAOYSA-N | $1.4\times10^4$<br>$5.4\times10^3$<br>$1.2\times10^3$ | | Wang et al. (2017)<br>Wang et al. (2017)<br>Wang et al. (2017) | Q<br>Q<br>Q | 80, 238<br>80, 239<br>80, 240 |
| MCM:C86OOH<br>$C_8H_{14}O_4$<br>DHFNMJQLYJIWHS-UHFFFAOYSA-N | $1.4\times10^6$<br>$3.0\times10^5$<br>$3.0\times10^3$ | | Wang et al. (2017)<br>Wang et al. (2017)<br>Wang et al. (2017) | Q<br>Q<br>Q | 80, 238<br>80, 239<br>80, 240 |
| MCM:C87CO<br>$C_8H_{10}O_4$<br>ZGSYEZWRFTZRGE-UHFFFAOYSA-N | $1.5\times10^7$<br>$7.1\times10^4$<br>$2.2\times10^1$ | | Wang et al. (2017)<br>Wang et al. (2017)<br>Wang et al. (2017) | Q<br>Q<br>Q | 80, 238<br>80, 239<br>80, 240 |
| MCM:C87OOH<br>$C_8H_{12}O_5$<br>ZOZQVMIZJFBYCH-UHFFFAOYSA-N | $1.2\times10^9$<br>$2.8\times10^6$<br>$9.3\times10^2$ | | Wang et al. (2017)<br>Wang et al. (2017)<br>Wang et al. (2017) | Q<br>Q<br>Q | 80, 238<br>80, 239<br>80, 240 |
| MCM:C8CO2DBAL<br>$C_8H_{10}O_3$<br>WFUAEUZWLUGFPU-UHFFFAOYSA-N | $6.8\times10^4$<br>$3.9\times10^4$<br>$2.2$ | | Wang et al. (2017)<br>Wang et al. (2017)<br>Wang et al. (2017) | Q<br>Q<br>Q | 80, 238<br>80, 239<br>80, 240 |
| MCM:C8CO2DCO3H<br>$C_8H_{10}O_5$<br>AFZRTBPBEHNFPV-UHFFFAOYSA-N | $7.4\times10^7$<br>$1.0\times10^6$<br>$1.7\times10^{-1}$ | | Wang et al. (2017)<br>Wang et al. (2017)<br>Wang et al. (2017) | Q<br>Q<br>Q | 80, 238<br>80, 239<br>80, 240 |
| MCM:C8CO4DB<br>$C_8H_8O_4$<br>XMBXSFIUGNBAMW-UHFFFAOYSA-N | $3.9\times10^7$<br>$1.1\times10^6$<br>$2.4$ | | Wang et al. (2017)<br>Wang et al. (2017)<br>Wang et al. (2017) | Q<br>Q<br>Q | 80, 238<br>80, 239<br>80, 240 |
| MCM:C8DCO2CO3H<br>$C_8H_{10}O_5$<br>LSJULIYFTAQYGE-UHFFFAOYSA-N | $8.3\times10^7$<br>$1.2\times10^6$<br>$1.9\times10^{-1}$ | | Wang et al. (2017)<br>Wang et al. (2017)<br>Wang et al. (2017) | Q<br>Q<br>Q | 80, 238<br>80, 239<br>80, 240 |
| MCM:CO4DBC8<br>$C_8H_8O_4$<br>UUVQUVHMBIAKFN-UHFFFAOYSA-N | $3.9\times10^7$<br>$1.1\times10^6$<br>$1.0$ | | Wang et al. (2017)<br>Wang et al. (2017)<br>Wang et al. (2017) | Q<br>Q<br>Q | 80, 238<br>80, 239<br>80, 240 |
| MCM:MC7CO2OOH<br>$C_8H_{12}O_4$<br>XEFYLFAPQOHXQI-UHFFFAOYSA-N | $5.4\times10^6$<br>$3.0\times10^3$<br>$9.1\times10^3$ | | Wang et al. (2017)<br>Wang et al. (2017)<br>Wang et al. (2017) | Q<br>Q<br>Q | 80, 238<br>80, 239<br>80, 240 |
| MCM:C6CO4MEDB<br>$C_9H_{10}O_4$<br>KWNUZJFJQSMOTR-UHFFFAOYSA-N | $3.0\times10^7$<br>$1.1\times10^6$<br>$6.9\times10^{-1}$ | | Wang et al. (2017)<br>Wang et al. (2017)<br>Wang et al. (2017) | Q<br>Q<br>Q | 80, 238<br>80, 239<br>80, 240 |
| MCM:C6IPRCO4DB<br>$C_9H_{10}O_4$<br>PHOIWTKCTUWXFI-UHFFFAOYSA-N | $4.2\times10^7$<br>$8.7\times10^5$<br>$1.9\times10^{-1}$ | | Wang et al. (2017)<br>Wang et al. (2017)<br>Wang et al. (2017) | Q<br>Q<br>Q | 80, 238<br>80, 239<br>80, 240 |





Table A3.6: Ketones (RCOR) (...continued)

| Substance Formula (Trivial Name) [CAS Registry Number] InChIKey | $H_s^{cp}$ (at $T^\ominus$) $\left[\dfrac{\text{mol}}{\text{m}^3\,\text{Pa}}\right]$ | $\dfrac{\text{d}\ln H_s^{cp}}{\text{d}(1/T)}$ [K] | Reference | Type | Note |
|---|---|---|---|---|---|
| MCM:C6PRCO4DB $C_9H_{10}O_4$ ZAOQHXFQWRKJIE-UHFFFAOYSA-N | $3.7\times10^7$ $9.6\times10^5$ $1.9\times10^{-1}$ | | Wang et al. (2017) Wang et al. (2017) Wang et al. (2017) | Q Q Q | 80, 238 80, 239 80, 240 |
| MCM:C7CO4EDB $C_9H_{10}O_4$ ZLFZFHOVUMHREF-UHFFFAOYSA-N | $3.0\times10^7$ $1.3\times10^6$ $1.0$ | | Wang et al. (2017) Wang et al. (2017) Wang et al. (2017) | Q Q Q | 80, 238 80, 239 80, 240 |
| MCM:C7ECO4DB $C_9H_{10}O_4$ AWHQTSPUFRSDOO-UHFFFAOYSA-N | $3.0\times10^7$ $1.3\times10^6$ $7.3\times10^{-1}$ | | Wang et al. (2017) Wang et al. (2017) Wang et al. (2017) | Q Q Q | 80, 238 80, 239 80, 240 |
| MCM:C827CHO $C_9H_{14}O_3$ WRGXNTJIWBXAOX-UHFFFAOYSA-N | $9.6\times10^3$ $8.5\times10^3$ $3.2\times10^2$ | | Wang et al. (2017) Wang et al. (2017) Wang et al. (2017) | Q Q Q | 80, 238 80, 239 80, 240 |
| MCM:C828CHO $C_9H_{12}O_4$ YZUHPZZEPOWVFM-UHFFFAOYSA-N | $7.1\times10^6$ $5.8\times10^4$ $1.3\times10^2$ | | Wang et al. (2017) Wang et al. (2017) Wang et al. (2017) | Q Q Q | 80, 238 80, 239 80, 240 |
| MCM:C87CO3H $C_9H_{12}O_6$ AJHPLULEOIFIGV-UHFFFAOYSA-N | $1.5\times10^{10}$ $5.3\times10^7$ $2.8\times10^3$ | | Wang et al. (2017) Wang et al. (2017) Wang et al. (2017) | Q Q Q | 80, 238 80, 239 80, 240 |
| MCM:C88CHO $C_9H_{12}O_3$ NXCXUGLKKPDWEC-UHFFFAOYSA-N | $3.5\times10^4$ $8.9\times10^4$ $8.5\times10^3$ | | Wang et al. (2017) Wang et al. (2017) Wang et al. (2017) | Q Q Q | 80, 238 80, 239 80, 240 |
| MCM:C914CO $C_9H_{10}O_4$ RDBINALOBQAVHT-UHFFFAOYSA-N | $2.3\times10^7$ $1.5\times10^7$ $5.8\times10^4$ | | Wang et al. (2017) Wang et al. (2017) Wang et al. (2017) | Q Q Q | 80, 238 80, 239 80, 240 |
| MCM:C914OOH $C_9H_{12}O_5$ ZERYOMDBUHTQAV-UHFFFAOYSA-N | $2.6\times10^9$ $1.6\times10^8$ $3.0\times10^4$ | | Wang et al. (2017) Wang et al. (2017) Wang et al. (2017) | Q Q Q | 80, 238 80, 239 80, 240 |
| MCM:C915OOH $C_9H_{14}O_4$ IHUGTHJLMVJZBZ-UHFFFAOYSA-N | $4.0\times10^6$ $4.2\times10^6$ $2.8\times10^5$ | | Wang et al. (2017) Wang et al. (2017) Wang et al. (2017) | Q Q Q | 80, 238 80, 239 80, 240 |
| MCM:C916OOH $C_9H_{14}O_5$ FURCCROLNKVFJP-UHFFFAOYSA-N | $1.0\times10^9$ $4.8\times10^7$ $4.9\times10^3$ | | Wang et al. (2017) Wang et al. (2017) Wang et al. (2017) | Q Q Q | 80, 238 80, 239 80, 240 |
| MCM:C918OOH $C_9H_{14}O_4$ RVRGXPTVZZTPQO-UHFFFAOYSA-N | $4.0\times10^6$ $7.8\times10^5$ $1.1\times10^5$ | | Wang et al. (2017) Wang et al. (2017) Wang et al. (2017) | Q Q Q | 80, 238 80, 239 80, 240 |
| MCM:C919OOH $C_9H_{14}O_5$ HGDKSMVLGYFVMB-UHFFFAOYSA-N | $1.0\times10^9$ $4.5\times10^7$ $2.6\times10^4$ | | Wang et al. (2017) Wang et al. (2017) Wang et al. (2017) | Q Q Q | 80, 238 80, 239 80, 240 |





Table A3.6: Ketones (RCOR) (. . . continued)

| Substance Formula (Trivial Name) [CAS Registry Number] InChIKey | $H_s^{cp}$ (at $T^\ominus$) $\left[\dfrac{\text{mol}}{\text{m}^3\,\text{Pa}}\right]$ | $\dfrac{\text{d}\ln H_s^{cp}}{\text{d}(1/T)}$ [K] | Reference | Type | Note |
|---|---|---|---|---|---|
| MCM:C926OOH | $7.3\times10^8$ | | Wang et al. (2017) | Q | 80, 238 |
| $C_9H_{14}O_5$ | $5.5\times10^6$ | | Wang et al. (2017) | Q | 80, 239 |
| YGLJXDYJGXWZRQ-UHFFFAOYSA-N | $2.0\times10^4$ | | Wang et al. (2017) | Q | 80, 240 |
| MCM:C930OOH | $7.3\times10^8$ | | Wang et al. (2017) | Q | 80, 238 |
| $C_9H_{14}O_5$ | $2.6\times10^7$ | | Wang et al. (2017) | Q | 80, 239 |
| UQPOFUKCKANZIY-UHFFFAOYSA-N | $2.0\times10^3$ | | Wang et al. (2017) | Q | 80, 240 |
| MCM:CO4DBC9 | $2.5\times10^7$ | | Wang et al. (2017) | Q | 80, 238 |
| $C_9H_{10}O_4$ | $1.2\times10^6$ | | Wang et al. (2017) | Q | 80, 239 |
| KCIAXTRCVKQJBW-UHFFFAOYSA-N | $7.1$ | | Wang et al. (2017) | Q | 80, 240 |
| MCM:LMLKACO | $1.0\times10^7$ | | Wang et al. (2017) | Q | 80, 238 |
| $C_9H_{12}O_4$ | $8.1\times10^5$ | | Wang et al. (2017) | Q | 80, 239 |
| JZUAMOXWRDMCCH-UHFFFAOYSA-N | $8.9\times10^3$ | | Wang et al. (2017) | Q | 80, 240 |
| MCM:LMLKAOOH | $1.2\times10^9$ | | Wang et al. (2017) | Q | 80, 238 |
| $C_9H_{14}O_5$ | $1.7\times10^7$ | | Wang et al. (2017) | Q | 80, 239 |
| VTXNRXDJNZTEJV-UHFFFAOYSA-N | $9.1\times10^5$ | | Wang et al. (2017) | Q | 80, 240 |
| MCM:LMLKBCO | $1.3\times10^7$ | | Wang et al. (2017) | Q | 80, 238 |
| $C_9H_{12}O_4$ | $5.5\times10^5$ | | Wang et al. (2017) | Q | 80, 239 |
| UMGHNOQMPCKCSD-UHFFFAOYSA-N | $6.5\times10^3$ | | Wang et al. (2017) | Q | 80, 240 |
| MCM:LMLKBOOH | $1.0\times10^9$ | | Wang et al. (2017) | Q | 80, 238 |
| $C_9H_{14}O_5$ | $3.6\times10^7$ | | Wang et al. (2017) | Q | 80, 239 |
| YVDVZIZUYUAKKJ-UHFFFAOYSA-N | $3.0\times10^6$ | | Wang et al. (2017) | Q | 80, 240 |
| MCM:LMLKET | $1.1\times10^6$ | 13000 | Wieser et al. (2023) | Q | 437 |
| $C_9H_{14}O_3$ | $1.4\times10^4$ | | Wang et al. (2017) | Q | 80, 238 |
| CWEQHJLFCKMWEE-UHFFFAOYSA-N | $5.4\times10^4$ | | Wang et al. (2017) | Q | 80, 239 |
| | $3.8\times10^4$ | | Wang et al. (2017) | Q | 80, 240 |
| MCM:NORLIMAL | $6.1\times10^2$ | 11000 | Wieser et al. (2023) | Q | 437 |
| $C_9H_{14}O_2$ | $4.9\times10^1$ | | Wang et al. (2017) | Q | 80, 238 |
| WPNOYHDEEVPEJH-UHFFFAOYSA-N | $1.0\times10^2$ | | Wang et al. (2017) | Q | 80, 239 |
| | $9.8\times10^1$ | | Wang et al. (2017) | Q | 80, 240 |
| MCM:NORPINAL | $5.3\times10^1$ | | Wang et al. (2017) | Q | 80, 238 |
| $C_9H_{14}O_2$ | $2.2\times10^2$ | | Wang et al. (2017) | Q | 80, 239 |
| CRBGTXDFEDFKSU-UHFFFAOYSA-N | $4.7\times10^2$ | | Wang et al. (2017) | Q | 80, 240 |
| MCM:C1010OOH | $2.0\times10^6$ | | Wang et al. (2017) | Q | 80, 238 |
| $C_{10}H_{16}O_4$ | $3.8\times10^5$ | | Wang et al. (2017) | Q | 80, 239 |
| PQCOZMUQYGSZBV-UHFFFAOYSA-N | $1.3\times10^4$ | | Wang et al. (2017) | Q | 80, 240 |
| MCM:C1012CO | $7.4\times10^3$ | | Wang et al. (2017) | Q | 80, 238 |
| $C_{10}H_{16}O_3$ | $6.5\times10^4$ | | Wang et al. (2017) | Q | 80, 239 |
| ZTXNVHZVZBDWRG-UHFFFAOYSA-N | $1.4\times10^4$ | | Wang et al. (2017) | Q | 80, 240 |
| MCM:C1012OOH | $9.8\times10^5$ | | Wang et al. (2017) | Q | 80, 238 |
| $C_{10}H_{18}O_4$ | $3.5\times10^5$ | | Wang et al. (2017) | Q | 80, 239 |
| NQAGNHJKHGEOIP-UHFFFAOYSA-N | $1.2\times10^3$ | | Wang et al. (2017) | Q | 80, 240 |





Table A3.6: Ketones (RCOR) (...continued)

| Substance Formula (Trivial Name) [CAS Registry Number] InChIKey | $H_s^{cp}$ (at $T^\ominus$) $\left[\dfrac{\mathrm{mol}}{\mathrm{m^3\,Pa}}\right]$ | $\dfrac{\mathrm{d}\ln H_s^{cp}}{\mathrm{d}(1/T)}$ [K] | Reference | Type | Note |
|---|---|---|---|---|---|
| MCM:C106OOH | $6.8\times10^8$ | | Wang et al. (2017) | Q | 80, 238 |
| $C_{10}H_{16}O_5$ | $3.2\times10^7$ | | Wang et al. (2017) | Q | 80, 239 |
| QENSMPURVPBEBQ-UHFFFAOYSA-N | $1.8\times10^5$ | | Wang et al. (2017) | Q | 80, 240 |
| MCM:C107OOH | $2.1\times10^6$ | | Wang et al. (2017) | Q | 80, 238 |
| $C_{10}H_{16}O_4$ | $1.6\times10^4$ | | Wang et al. (2017) | Q | 80, 239 |
| JYIMELQJWXRSLA-UHFFFAOYSA-N | $3.2\times10^5$ | | Wang et al. (2017) | Q | 80, 240 |
| MCM:C108OOH | $6.8\times10^8$ | | Wang et al. (2017) | Q | 80, 238 |
| $C_{10}H_{16}O_5$ | $1.7\times10^7$ | | Wang et al. (2017) | Q | 80, 239 |
| AIXQDYAMZUITKH-UHFFFAOYSA-N | $2.2\times10^3$ | | Wang et al. (2017) | Q | 80, 240 |
| MCM:C109CO | $3.9\times10^4$ | | Wang et al. (2017) | Q | 80, 238 |
| $C_{10}H_{14}O_3$ | $5.6\times10^3$ | | Wang et al. (2017) | Q | 80, 239 |
| GPWOGFGLGKGEOT-UHFFFAOYSA-N | $8.9\times10^1$ | | Wang et al. (2017) | Q | 80, 240 |
| MCM:C109OOH | $3.2\times10^6$ | | Wang et al. (2017) | Q | 80, 238 |
| $C_{10}H_{16}O_4$ | $5.8\times10^4$ | | Wang et al. (2017) | Q | 80, 239 |
| ODTCGEGBAGAWAJ-UHFFFAOYSA-N | $5.6\times10^4$ | | Wang et al. (2017) | Q | 80, 240 |
| MCM:LIMALACO | $4.3\times10^4$ | 13000 | Wieser et al. (2023) | Q | 437 |
| $C_{10}H_{14}O_3$ | $2.6\times10^4$ | | Wang et al. (2017) | Q | 80, 238 |
| OXUPJDFJWCMZMO-UHFFFAOYSA-N | $2.9\times10^3$ | | Wang et al. (2017) | Q | 80, 239 |
| | $3.2\times10^1$ | | Wang et al. (2017) | Q | 80, 240 |
| MCM:LIMALAOOH | $2.5\times10^5$ | 15000 | Wieser et al. (2023) | Q | 437 |
| $C_{10}H_{16}O_4$ | $3.5\times10^6$ | | Wang et al. (2017) | Q | 80, 238 |
| DVORBFWFCUOYLW-UHFFFAOYSA-N | $1.1\times10^4$ | | Wang et al. (2017) | Q | 80, 239 |
| | $4.2\times10^3$ | | Wang et al. (2017) | Q | 80, 240 |
| MCM:LIMALBCO | $3.5\times10^4$ | | Wang et al. (2017) | Q | 80, 238 |
| $C_{10}H_{14}O_3$ | $2.0\times10^3$ | | Wang et al. (2017) | Q | 80, 239 |
| OFCYPGCYWHDSNB-UHFFFAOYSA-N | $8.1\times10^1$ | | Wang et al. (2017) | Q | 80, 240 |
| MCM:LIMALBOOH | $2.4\times10^5$ | 15000 | Wieser et al. (2023) | Q | 437 |
| $C_{10}H_{16}O_4$ | $2.9\times10^6$ | | Wang et al. (2017) | Q | 80, 238 |
| DCWLKDSHBWMMQU-UHFFFAOYSA-N | $2.0\times10^4$ | | Wang et al. (2017) | Q | 80, 239 |
| | $1.4\times10^4$ | | Wang et al. (2017) | Q | 80, 240 |
| MCM:LIMAL | $4.0\times10^2$ | 11000 | Wieser et al. (2023) | Q | 437 |
| $C_{10}H_{16}O_2$ | $3.8\times10^1$ | | Wang et al. (2017) | Q | 80, 238 |
| OGCGCISRMFSLTC-UHFFFAOYSA-N | $7.4\times10^1$ | | Wang et al. (2017) | Q | 80, 239 |
| | $2.1\times10^2$ | | Wang et al. (2017) | Q | 80, 240 |
| MCM:PINAL | $4.3\times10^1$ | | Wang et al. (2017) | Q | 80, 238 |
| $C_{10}H_{16}O_2$ | $2.0\times10^2$ | | Wang et al. (2017) | Q | 80, 239 |
| GCHDWVBHKDJOKU-UHFFFAOYSA-N | $1.7\times10^3$ | | Wang et al. (2017) | Q | 80, 240 |
| MCM:PINALOOH | $2.1\times10^6$ | | Wang et al. (2017) | Q | 80, 238 |
| $C_{10}H_{16}O_4$ | $1.0\times10^6$ | | Wang et al. (2017) | Q | 80, 239 |
| IKYOAMQPSDRUHL-UHFFFAOYSA-N | $5.5\times10^4$ | | Wang et al. (2017) | Q | 80, 240 |





Table A3.6: Ketones (RCOR) (...continued)

| Substance Formula (Trivial Name) [CAS Registry Number] InChIKey | $H_s^{cp}$ (at $T^{\ominus}$) $\left[\dfrac{\text{mol}}{\text{m}^3\,\text{Pa}}\right]$ | $\dfrac{\text{d}\ln H_s^{cp}}{\text{d}(1/T)}$ [K] | Reference | Type | Note |
|---|---|---|---|---|---|
| MCM:C116CO | $3.0 \times 10^4$ | | Wang et al. (2017) | Q | 80, 238 |
| $C_{11}H_{16}O_3$ | $2.0 \times 10^4$ | | Wang et al. (2017) | Q | 80, 239 |
| XGVKCXDPIYHMTH-UHFFFAOYSA-N | $1.1 \times 10^3$ | | Wang et al. (2017) | Q | 80, 240 |
| MCM:C116OOH | $2.6 \times 10^6$ | | Wang et al. (2017) | Q | 80, 238 |
| $C_{11}H_{18}O_4$ | $2.0 \times 10^6$ | | Wang et al. (2017) | Q | 80, 239 |
| GLQKGYMROFUHSE-UHFFFAOYSA-N | $2.7 \times 10^4$ | | Wang et al. (2017) | Q | 80, 240 |
| MCM:C116CHO | $2.8 \times 10^4$ | | Wang et al. (2017) | Q | 80, 238 |
| $C_{12}H_{18}O_3$ | $1.4 \times 10^5$ | | Wang et al. (2017) | Q | 80, 239 |
| LCDZSXGLJAAQMI-UHFFFAOYSA-N | $1.6 \times 10^3$ | | Wang et al. (2017) | Q | 80, 240 |
| MCM:C116CO3H | $3.0 \times 10^7$ | | Wang et al. (2017) | Q | 80, 238 |
| $C_{12}H_{18}O_5$ | $2.4 \times 10^6$ | | Wang et al. (2017) | Q | 80, 239 |
| DYOBAPPYFCSOAS-UHFFFAOYSA-N | $2.0 \times 10^3$ | | Wang et al. (2017) | Q | 80, 240 |
| MCM:C1210OOH | $2.2 \times 10^6$ | | Wang et al. (2017) | Q | 80, 238 |
| $C_{12}H_{20}O_4$ | $4.7 \times 10^6$ | | Wang et al. (2017) | Q | 80, 239 |
| KZRFFJDVJJJBDC-UHFFFAOYSA-N | $4.3 \times 10^3$ | | Wang et al. (2017) | Q | 80, 240 |
| MCM:C129CO | $1.9 \times 10^7$ | | Wang et al. (2017) | Q | 80, 238 |
| $C_{12}H_{16}O_4$ | $8.7 \times 10^5$ | | Wang et al. (2017) | Q | 80, 239 |
| RGVZWLUABXGGBI-UHFFFAOYSA-N | $1.1 \times 10^3$ | | Wang et al. (2017) | Q | 80, 240 |
| MCM:C129OOH | $2.2 \times 10^9$ | | Wang et al. (2017) | Q | 80, 238 |
| $C_{12}H_{18}O_5$ | $1.5 \times 10^7$ | | Wang et al. (2017) | Q | 80, 239 |
| OVAMHHASBHCMIZ-UHFFFAOYSA-N | $2.2 \times 10^4$ | | Wang et al. (2017) | Q | 80, 240 |
| MCM:C1210CO3H | $2.8 \times 10^7$ | | Wang et al. (2017) | Q | 80, 238 |
| $C_{13}H_{20}O_5$ | $1.9 \times 10^6$ | | Wang et al. (2017) | Q | 80, 239 |
| BXGKTBFSJURUHS-UHFFFAOYSA-N | $6.0 \times 10^3$ | | Wang et al. (2017) | Q | 80, 240 |
| MCM:C131CO | $1.8 \times 10^4$ | | Wang et al. (2017) | Q | 80, 238 |
| $C_{13}H_{20}O_3$ | $2.1 \times 10^4$ | | Wang et al. (2017) | Q | 80, 239 |
| FDPPFMUIMWYEIX-UHFFFAOYSA-N | $3.0 \times 10^4$ | | Wang et al. (2017) | Q | 80, 240 |
| MCM:BCLKACO | $1.0 \times 10^7$ | | Wang et al. (2017) | Q | 80, 238 |
| $C_{14}H_{20}O_4$ | $3.2 \times 10^6$ | | Wang et al. (2017) | Q | 80, 239 |
| ARRZIWJBJSIOKE-UHFFFAOYSA-N | $4.0 \times 10^3$ | | Wang et al. (2017) | Q | 80, 240 |
| MCM:BCLKAOOH | $1.1 \times 10^9$ | | Wang et al. (2017) | Q | 80, 238 |
| $C_{14}H_{22}O_5$ | $1.7 \times 10^7$ | | Wang et al. (2017) | Q | 80, 239 |
| CKBNEDHFWOQWQD-UHFFFAOYSA-N | $2.6 \times 10^5$ | | Wang et al. (2017) | Q | 80, 240 |
| MCM:BCLKBCO | $1.4 \times 10^7$ | | Wang et al. (2017) | Q | 80, 238 |
| $C_{14}H_{20}O_4$ | $2.9 \times 10^6$ | | Wang et al. (2017) | Q | 80, 239 |
| NZHCLRCYZNBCMK-UHFFFAOYSA-N | $1.0 \times 10^4$ | | Wang et al. (2017) | Q | 80, 240 |
| MCM:BCLKBOOH | $1.1 \times 10^9$ | | Wang et al. (2017) | Q | 80, 238 |
| $C_{14}H_{22}O_5$ | $4.6 \times 10^7$ | | Wang et al. (2017) | Q | 80, 239 |
| CNYMBWKCKYTSRZ-UHFFFAOYSA-N | $1.5 \times 10^6$ | | Wang et al. (2017) | Q | 80, 240 |





Table A3.6: Ketones (RCOR) (...continued)

| Substance<br>Formula<br>(Trivial Name)<br>[CAS Registry Number]<br>InChIKey | $H_s^{cp}$<br>(at $T^{\ominus}$)<br>$\left[\dfrac{\text{mol}}{\text{m}^3\,\text{Pa}}\right]$ | $\dfrac{\text{d}\ln H_s^{cp}}{\text{d}(1/T)}$<br><br>[K] | Reference | Type | Note |
|---|---|---|---|---|---|
| MCM:BCLKCCO | $1.0\times10^7$ | | Wang et al. (2017) | Q | 80, 238 |
| $C_{14}H_{20}O_4$ | $1.2\times10^6$ | | Wang et al. (2017) | Q | 80, 239 |
| RVZWWCAHYGTHHX-UHFFFAOYSA-N | $1.5\times10^4$ | | Wang et al. (2017) | Q | 80, 240 |
| MCM:BCLKCOOH | $1.1\times10^9$ | | Wang et al. (2017) | Q | 80, 238 |
| $C_{14}H_{22}O_5$ | $1.5\times10^7$ | | Wang et al. (2017) | Q | 80, 239 |
| OVPKMRRDRUAGIL-UHFFFAOYSA-N | $6.6\times10^4$ | | Wang et al. (2017) | Q | 80, 240 |
| MCM:BCLKET | $1.5\times10^4$ | | Wang et al. (2017) | Q | 80, 238 |
| $C_{14}H_{22}O_3$ | $1.4\times10^5$ | | Wang et al. (2017) | Q | 80, 239 |
| UGDRYISIDBCIAI-UHFFFAOYSA-N | $1.0\times10^5$ | | Wang et al. (2017) | Q | 80, 240 |
| MCM:C141CO | $4.7\times10^1$ | | Wang et al. (2017) | Q | 80, 238 |
| $C_{14}H_{22}O_2$ | $1.6\times10^2$ | | Wang et al. (2017) | Q | 80, 239 |
| LUPUYUXNGLIEMW-UHFFFAOYSA-N | $7.1\times10^2$ | | Wang et al. (2017) | Q | 80, 240 |
| MCM:C146CO | $1.0\times10^7$ | | Wang et al. (2017) | Q | 80, 238 |
| $C_{14}H_{20}O_4$ | $1.1\times10^6$ | | Wang et al. (2017) | Q | 80, 239 |
| ODTZTSYYOIXMEF-UHFFFAOYSA-N | $1.0\times10^4$ | | Wang et al. (2017) | Q | 80, 240 |
| MCM:C146OOH | $1.1\times10^9$ | | Wang et al. (2017) | Q | 80, 238 |
| $C_{14}H_{22}O_5$ | $1.3\times10^7$ | | Wang et al. (2017) | Q | 80, 239 |
| SPXXDEIFDYFSMG-UHFFFAOYSA-N | $1.6\times10^5$ | | Wang et al. (2017) | Q | 80, 240 |
| MCM:BCALACO | $2.8\times10^4$ | | Wang et al. (2017) | Q | 80, 238 |
| $C_{15}H_{22}O_3$ | $3.6\times10^3$ | | Wang et al. (2017) | Q | 80, 239 |
| ULXUQSOGQJAAAV-UHFFFAOYSA-N | $4.7\times10^2$ | | Wang et al. (2017) | Q | 80, 240 |
| MCM:BCALAOOH | $3.2\times10^6$ | | Wang et al. (2017) | Q | 80, 238 |
| $C_{15}H_{24}O_4$ | $1.4\times10^4$ | | Wang et al. (2017) | Q | 80, 239 |
| CATQWYAHYHBBFL-UHFFFAOYSA-N | $1.8\times10^4$ | | Wang et al. (2017) | Q | 80, 240 |
| MCM:BCALBCO | $3.4\times10^4$ | | Wang et al. (2017) | Q | 80, 238 |
| $C_{15}H_{22}O_3$ | $2.9\times10^3$ | | Wang et al. (2017) | Q | 80, 239 |
| DVBAGSYXLVWTHM-UHFFFAOYSA-N | $3.6\times10^2$ | | Wang et al. (2017) | Q | 80, 240 |
| MCM:BCALBOOH | $2.8\times10^6$ | | Wang et al. (2017) | Q | 80, 238 |
| $C_{15}H_{24}O_4$ | $3.6\times10^4$ | | Wang et al. (2017) | Q | 80, 239 |
| BXDTVEFUTYSWEH-UHFFFAOYSA-N | $1.8\times10^5$ | | Wang et al. (2017) | Q | 80, 240 |
| MCM:BCAL | $3.7\times10^1$ | | Wang et al. (2017) | Q | 80, 238 |
| $C_{15}H_{24}O_2$ | $1.1\times10^2$ | | Wang et al. (2017) | Q | 80, 239 |
| PJJCDGMFEIUQRZ-UHFFFAOYSA-N | $3.2\times10^3$ | | Wang et al. (2017) | Q | 80, 240 |
| MCM:BCALCCO | $2.8\times10^4$ | | Wang et al. (2017) | Q | 80, 238 |
| $C_{15}H_{22}O_3$ | $4.7\times10^3$ | | Wang et al. (2017) | Q | 80, 239 |
| RMGPJKLPPRXORI-UHFFFAOYSA-N | $2.2\times10^2$ | | Wang et al. (2017) | Q | 80, 240 |
| MCM:BCALCOOH | $3.2\times10^6$ | | Wang et al. (2017) | Q | 80, 238 |
| $C_{15}H_{24}O_4$ | $1.7\times10^4$ | | Wang et al. (2017) | Q | 80, 239 |
| WQOUANLPJNTVBV-UHFFFAOYSA-N | $2.5\times10^4$ | | Wang et al. (2017) | Q | 80, 240 |





Table A3.6: Ketones (RCOR) (...continued)

| Substance Formula (Trivial Name) [CAS Registry Number] InChIKey | $H_s^{cp}$ (at $T^\ominus$) $\left[\dfrac{\text{mol}}{\text{m}^3\,\text{Pa}}\right]$ | $\dfrac{\text{d}\ln H_s^{cp}}{\text{d}(1/T)}$ [K] | Reference | Type | Note |
|---|---|---|---|---|---|
| MCM:PHGLYOX $C_8H_6O_2$ OJUGVDODNPJEEC-UHFFFAOYSA-N | $1.9\times10^3$ $8.9\times10^1$ $8.1\times10^{-1}$ | | Wang et al. (2017) Wang et al. (2017) Wang et al. (2017) | Q Q Q | 80, 238 80, 239 80, 240 |
| MCM:MPHGLYOX $C_9H_8O_2$ ILRFLXHICGHRIN-UHFFFAOYSA-N | $1.1\times10^3$ $1.1\times10^2$ $1.0$ | | Wang et al. (2017) Wang et al. (2017) Wang et al. (2017) | Q Q Q | 80, 238 80, 239 80, 240 |
| MCM:DMPHGLYOX $C_{10}H_{10}O_2$ COKLAMNWIKZGIN-UHFFFAOYSA-N | $6.5\times10^2$ $6.6\times10^1$ $1.0$ | | Wang et al. (2017) Wang et al. (2017) Wang et al. (2017) | Q Q Q | 80, 238 80, 239 80, 240 |
| MCM:EMPHGLYOX $C_{11}H_{12}O_2$ LRURTQBJFCJOFP-UHFFFAOYSA-N | $5.9\times10^2$ $5.3\times10^1$ $8.7\times10^{-1}$ | | Wang et al. (2017) Wang et al. (2017) Wang et al. (2017) | Q Q Q | 80, 238 80, 239 80, 240 |
| MCM:HOCH2COCHO $C_3H_4O_3$ JLPAWRLRMTZCSF-UHFFFAOYSA-N | $7.6\times10^3$ $4.4\times10^3$ $1.3$ | | Wang et al. (2017) Wang et al. (2017) Wang et al. (2017) | Q Q Q | 80, 238 80, 239 80, 240 |
| MCM:CCOCOCOH $C_4H_6O_3$ AWMGPEHDHJUMEN-UHFFFAOYSA-N | $7.1\times10^3$ $3.2\times10^3$ $1.4$ | | Wang et al. (2017) Wang et al. (2017) Wang et al. (2017) | Q Q Q | 80, 238 80, 239 80, 240 |
| MCM:CO13C4OH $C_4H_6O_3$ KATSYVMPKZTIRW-UHFFFAOYSA-N | $6.2\times10^3$ $2.3\times10^4$ $4.4\times10^2$ | | Wang et al. (2017) Wang et al. (2017) Wang et al. (2017) | Q Q Q | 80, 238 80, 239 80, 240 |
| MCM:CO2H3CHO $C_4H_6O_3$ GTYVZUSWKGYETP-UHFFFAOYSA-N | $1.6\times10^3$ $1.1\times10^4$ $1.0\times10^1$ | | Wang et al. (2017) Wang et al. (2017) Wang et al. (2017) | Q Q Q | 80, 238 80, 239 80, 240 |
| MCM:H13CO2CHO $C_4H_6O_4$ RAPYKXPBVBNJGC-UHFFFAOYSA-N | $1.0\times10^6$ $4.6\times10^5$ $3.8\times10^2$ | | Wang et al. (2017) Wang et al. (2017) Wang et al. (2017) | Q Q Q | 80, 238 80, 239 80, 240 |
| MCM:H1CO23CHO $C_4H_4O_4$ XSEYITBSKRUSNL-UHFFFAOYSA-N | $4.7\times10^6$ $1.2\times10^6$ $3.3\times10^1$ | | Wang et al. (2017) Wang et al. (2017) Wang et al. (2017) | Q Q Q | 80, 238 80, 239 80, 240 |
| MCM:HO1CO3CHO $C_4H_6O_3$ CUSSNCHZLYDUPJ-UHFFFAOYSA-N | $1.2\times10^5$ $4.0\times10^4$ $6.9\times10^1$ | | Wang et al. (2017) Wang et al. (2017) Wang et al. (2017) | Q Q Q | 80, 238 80, 239 80, 240 |
| MCM:HOCOC4DIAL $C_4H_4O_4$ JWBVDBDNUQOADG-UHFFFAOYSA-N | $1.4\times10^6$ $5.3\times10^5$ $1.5\times10^1$ | | Wang et al. (2017) Wang et al. (2017) Wang et al. (2017) | Q Q Q | 80, 238 80, 239 80, 240 |
| MCM:C4MCO2OH $C_5H_8O_3$ YKNUMNAVGAMCFT-UHFFFAOYSA-N | $8.7\times10^2$ $3.2\times10^3$ $2.6$ | | Wang et al. (2017) Wang et al. (2017) Wang et al. (2017) | Q Q Q | 80, 238 80, 239 80, 240 |



Table A3.6: Ketones (RCOR) (...continued)

| Substance Formula (Trivial Name) [CAS Registry Number] InChIKey | $H_s^{cp}$ (at $T^\ominus$) $\left[\dfrac{\text{mol}}{\text{m}^3\,\text{Pa}}\right]$ | $\dfrac{\text{d}\ln H_s^{cp}}{\text{d}(1/T)}$ [K] | Reference | Type | Note |
|---|---|---|---|---|---|
| MCM:C512OH | $1.1\times10^5$ | | Wang et al. (2017) | Q | 80, 238 |
| $C_5H_8O_3$ | $1.5\times10^5$ | | Wang et al. (2017) | Q | 80, 239 |
| ZSJOVUGBPPFONP-UHFFFAOYSA-N | $5.6\times10^3$ | | Wang et al. (2017) | Q | 80, 240 |
| MCM:C5134CO2OH | $9.8\times10^5$ | | Wang et al. (2017) | Q | 80, 238 |
| $C_5H_6O_4$ | $4.2\times10^5$ | | Wang et al. (2017) | Q | 80, 239 |
| UFXCMYYHMGNRSQ-UHFFFAOYSA-N | $1.6\times10^1$ | | Wang et al. (2017) | Q | 80, 240 |
| MCM:C513CO | $7.1\times10^7$ | | Wang et al. (2017) | Q | 80, 238 |
| $C_5H_6O_4$ | $5.9\times10^6$ | | Wang et al. (2017) | Q | 80, 239 |
| GBFAZHBQOBSKET-UHFFFAOYSA-N | $3.6\times10^2$ | | Wang et al. (2017) | Q | 80, 240 |
| MCM:C513OH | $3.6\times10^6$ | | Wang et al. (2017) | Q | 80, 238 |
| $C_5H_8O_4$ | $8.9\times10^7$ | | Wang et al. (2017) | Q | 80, 239 |
| VLANIYKAIBIISW-UHFFFAOYSA-N | $4.6\times10^3$ | | Wang et al. (2017) | Q | 80, 240 |
| MCM:C513OOH | $8.9\times10^9$ | | Wang et al. (2017) | Q | 80, 238 |
| $C_5H_8O_5$ | $2.5\times10^8$ | | Wang et al. (2017) | Q | 80, 239 |
| FJRVRXOZJCMNIT-UHFFFAOYSA-N | $5.8\times10^4$ | | Wang et al. (2017) | Q | 80, 240 |
| MCM:C514CO23OH | $4.3\times10^5$ | | Wang et al. (2017) | Q | 80, 238 |
| $C_5H_8O_4$ | $4.5\times10^6$ | | Wang et al. (2017) | Q | 80, 239 |
| JLVNMCMGUWUGCO-UHFFFAOYSA-N | $4.3\times10^2$ | | Wang et al. (2017) | Q | 80, 240 |
| MCM:C520OH | $1.1\times10^6$ | | Wang et al. (2017) | Q | 80, 238 |
| $C_5H_8O_4$ | $1.1\times10^6$ | | Wang et al. (2017) | Q | 80, 239 |
| BXHDMMHJHUOSNS-UHFFFAOYSA-N | $1.7\times10^2$ | | Wang et al. (2017) | Q | 80, 240 |
| MCM:C520OOH | $7.1\times10^9$ | | Wang et al. (2017) | Q | 80, 238 |
| $C_5H_8O_5$ | $3.0\times10^6$ | | Wang et al. (2017) | Q | 80, 239 |
| BCWSZSNCYSGOQJ-UHFFFAOYSA-N | $2.9\times10^2$ | | Wang et al. (2017) | Q | 80, 240 |
| MCM:C5CO3OH | $9.8\times10^5$ | | Wang et al. (2017) | Q | 80, 238 |
| $C_5H_6O_4$ | $6.2\times10^5$ | | Wang et al. (2017) | Q | 80, 239 |
| UUUQEQKLOQQXBA-UHFFFAOYSA-N | $1.4\times10^1$ | | Wang et al. (2017) | Q | 80, 240 |
| MCM:C5DICAROOH | $6.2\times10^8$ | | Wang et al. (2017) | Q | 80, 238 |
| $C_5H_8O_5$ | $1.3\times10^7$ | | Wang et al. (2017) | Q | 80, 239 |
| SXCXUVJXNGWYKY-UHFFFAOYSA-N | $2.2\times10^4$ | | Wang et al. (2017) | Q | 80, 240 |
| MCM:CO3H4CHO | $1.3\times10^3$ | | Wang et al. (2017) | Q | 80, 238 |
| $C_5H_8O_3$ | $7.1\times10^3$ | | Wang et al. (2017) | Q | 80, 239 |
| GHHQRUFNLVFZPQ-UHFFFAOYSA-N | $5.9$ | | Wang et al. (2017) | Q | 80, 240 |
| MCM:H1C23C4CHO | $3.6\times10^6$ | | Wang et al. (2017) | Q | 80, 238 |
| $C_5H_6O_4$ | $1.1\times10^6$ | | Wang et al. (2017) | Q | 80, 239 |
| VBLXHDGHHJBCMK-UHFFFAOYSA-N | $3.3\times10^2$ | | Wang et al. (2017) | Q | 80, 240 |
| MCM:HMVKBCHO | $3.6\times10^5$ | 11000 | Wieser et al. (2023) | Q | 437 |
| $C_5H_8O_3$ | $1.1\times10^5$ | | Wang et al. (2017) | Q | 80, 238 |
| XYGGNMZYIGPNCC-UHFFFAOYSA-N | $1.3\times10^5$ | | Wang et al. (2017) | Q | 80, 239 |
| | $2.5\times10^2$ | | Wang et al. (2017) | Q | 80, 240 |



Table A3.6: Ketones (RCOR) (...continued)

| Substance<br>Formula<br>(Trivial Name)<br>[CAS Registry Number]<br>InChIKey | $H_s^{cp}$<br>(at $T^\ominus$)<br>$\left[\dfrac{\text{mol}}{\text{m}^3\,\text{Pa}}\right]$ | $\dfrac{\text{d}\ln H_s^{cp}}{\text{d}(1/T)}$<br><br>[K] | Reference | Type | Note |
|---|---|---|---|---|---|
| MCM:HO2CO4CHO<br>$C_5H_8O_3$<br>NSUJYXKDWVMPFQ-UHFFFAOYSA-N | $1.1\times10^5$<br>$2.5\times10^4$<br>$2.5\times10^1$ | | Wang et al. (2017)<br>Wang et al. (2017)<br>Wang et al. (2017) | Q<br>Q<br>Q | 80, 238<br>80, 239<br>80, 240 |
| MCM:HOIPRGLYOX<br>$C_5H_8O_3$<br>ROKGRLJNHSUEOC-UHFFFAOYSA-N | $1.1\times10^5$<br>$3.2\times10^4$<br>$1.9\times10^1$ | | Wang et al. (2017)<br>Wang et al. (2017)<br>Wang et al. (2017) | Q<br>Q<br>Q | 80, 238<br>80, 239<br>80, 240 |
| MCM:IEC2OOH<br>$C_5H_8O_5$<br>KGZJHQJCDPZOFG-UHFFFAOYSA-N | $3.0\times10^8$<br>$2.0\times10^6$<br>$2.3\times10^2$ | | Wang et al. (2017)<br>Wang et al. (2017)<br>Wang et al. (2017) | Q<br>Q<br>Q | 80, 238<br>80, 239<br>80, 240 |
| MCM:MBOCOCO<br>$C_5H_8O_3$<br>OCAJTNNCRSVJNU-UHFFFAOYSA-N | $3.9\times10^3$<br>$6.8\times10^2$<br>$8.3\times10^{-1}$ | | Wang et al. (2017)<br>Wang et al. (2017)<br>Wang et al. (2017) | Q<br>Q<br>Q | 80, 238<br>80, 239<br>80, 240 |
| MCM:C43OHCOCHO<br>$C_6H_{10}O_3$<br>NMZPCKZSNUOBSC-UHFFFAOYSA-N | $1.0\times10^5$<br>$2.8\times10^4$<br>$7.8$ | | Wang et al. (2017)<br>Wang et al. (2017)<br>Wang et al. (2017) | Q<br>Q<br>Q | 80, 238<br>80, 239<br>80, 240 |
| MCM:C517CHO<br>$C_6H_{10}O_3$<br>JGHPNZFTVGSLKF-UHFFFAOYSA-N | $9.5\times10^2$<br>$1.0\times10^5$<br>$7.8\times10^5$<br>$2.8\times10^3$ | 13000 | Wieser et al. (2023)<br>Wang et al. (2017)<br>Wang et al. (2017)<br>Wang et al. (2017) | Q<br>Q<br>Q<br>Q | 437<br>80, 238<br>80, 239<br>80, 240 |
| MCM:C519CHO<br>$C_6H_{10}O_3$<br>PVAHQKAVPKMXGV-UHFFFAOYSA-N | $2.8\times10^6$<br>$1.0\times10^5$<br>$2.6\times10^5$<br>$6.9\times10^3$ | 11000 | Wieser et al. (2023)<br>Wang et al. (2017)<br>Wang et al. (2017)<br>Wang et al. (2017) | Q<br>Q<br>Q<br>Q | 437<br>80, 238<br>80, 239<br>80, 240 |
| MCM:C5COOHCO3H<br>$C_6H_6O_6$<br>DETDOHNNOYOALO-UHFFFAOYSA-N | $4.7\times10^9$<br>$3.2\times10^9$<br>$8.5\times10^1$ | | Wang et al. (2017)<br>Wang et al. (2017)<br>Wang et al. (2017) | Q<br>Q<br>Q | 80, 238<br>80, 239<br>80, 240 |
| MCM:C5MO132OH<br>$C_6H_{10}O_3$<br>YVTXQDAOLHXGGS-UHFFFAOYSA-N | $1.2\times10^3$<br>$3.0\times10^3$<br>$2.7$ | | Wang et al. (2017)<br>Wang et al. (2017)<br>Wang et al. (2017) | Q<br>Q<br>Q | 80, 238<br>80, 239<br>80, 240 |
| MCM:C6134CO2OH<br>$C_6H_8O_4$<br>ZUSWTYLGBZLTMD-UHFFFAOYSA-N | $7.8\times10^5$<br>$1.8\times10^5$<br>$1.1\times10^1$ | | Wang et al. (2017)<br>Wang et al. (2017)<br>Wang et al. (2017) | Q<br>Q<br>Q | 80, 238<br>80, 239<br>80, 240 |
| MCM:C614CO23OH<br>$C_6H_{10}O_4$<br>FTHGMBBEYTXWQT-UHFFFAOYSA-N | $3.4\times10^5$<br>$3.6\times10^6$<br>$1.9\times10^2$ | | Wang et al. (2017)<br>Wang et al. (2017)<br>Wang et al. (2017) | Q<br>Q<br>Q | 80, 238<br>80, 239<br>80, 240 |
| MCM:C615CO2OH<br>$C_6H_8O_3$<br>AEWAJJBQFQHEAB-UHFFFAOYSA-N | $2.1\times10^4$<br>$1.1\times10^5$<br>$3.2\times10^3$ | | Wang et al. (2017)<br>Wang et al. (2017)<br>Wang et al. (2017) | Q<br>Q<br>Q | 80, 238<br>80, 239<br>80, 240 |
| MCM:C616OH<br>$C_6H_8O_4$<br>VJOSQGBCXQMKDE-UHFFFAOYSA-N | $9.3\times10^5$<br>$7.4\times10^6$<br>$8.9\times10^2$ | | Wang et al. (2017)<br>Wang et al. (2017)<br>Wang et al. (2017) | Q<br>Q<br>Q | 80, 238<br>80, 239<br>80, 240 |





Table A3.6: Ketones (RCOR) (...continued)

| Substance Formula (Trivial Name) [CAS Registry Number] InChIKey | $H_s^{cp}$ (at $T^\ominus$) $\left[\dfrac{\mathrm{mol}}{\mathrm{m^3\,Pa}}\right]$ | $\dfrac{\mathrm{d}\ln H_s^{cp}}{\mathrm{d}(1/T)}$ [K] | Reference | Type | Note |
|---|---|---|---|---|---|
| MCM:C617OH | $3.2\times10^3$ | | Wang et al. (2017) | Q | 80, 238 |
| $C_6H_{10}O_3$ | $4.8\times10^3$ | | Wang et al. (2017) | Q | 80, 239 |
| GSQCYSIEWGMUPC-UHFFFAOYSA-N | $7.6\times10^1$ | | Wang et al. (2017) | Q | 80, 240 |
| MCM:C620OH | $4.2\times10^6$ | | Wang et al. (2017) | Q | 80, 238 |
| $C_6H_8O_4$ | $9.6\times10^6$ | | Wang et al. (2017) | Q | 80, 239 |
| SMLWDBHBBFROKP-UHFFFAOYSA-N | $4.1\times10^3$ | | Wang et al. (2017) | Q | 80, 240 |
| MCM:C628OH | $4.0\times10^6$ | | Wang et al. (2017) | Q | 80, 238 |
| $C_6H_{10}O_4$ | $3.6\times10^7$ | | Wang et al. (2017) | Q | 80, 239 |
| XRZKKMNIWXFNJQ-UHFFFAOYSA-N | $3.6\times10^3$ | | Wang et al. (2017) | Q | 80, 240 |
| MCM:C628OOH | $6.3\times10^9$ | | Wang et al. (2017) | Q | 80, 238 |
| $C_6H_{10}O_5$ | $5.4\times10^7$ | | Wang et al. (2017) | Q | 80, 239 |
| BGLCDSUTRBZUMD-UHFFFAOYSA-N | $3.5\times10^3$ | | Wang et al. (2017) | Q | 80, 240 |
| MCM:C629OH | $2.0\times10^6$ | | Wang et al. (2017) | Q | 80, 238 |
| $C_6H_{10}O_4$ | $1.1\times10^7$ | | Wang et al. (2017) | Q | 80, 239 |
| MIXLBBAGKJLCSO-UHFFFAOYSA-N | $3.6\times10^3$ | | Wang et al. (2017) | Q | 80, 240 |
| MCM:C629OOH | $4.9\times10^9$ | | Wang et al. (2017) | Q | 80, 238 |
| $C_6H_{10}O_5$ | $2.0\times10^7$ | | Wang et al. (2017) | Q | 80, 239 |
| PARGUUHBZPMBOS-UHFFFAOYSA-N | $2.5\times10^4$ | | Wang et al. (2017) | Q | 80, 240 |
| MCM:C630CO | $2.0\times10^5$ | | Wang et al. (2017) | Q | 80, 238 |
| $C_6H_8O_3$ | $3.4\times10^5$ | | Wang et al. (2017) | Q | 80, 239 |
| AUMGTZPDEFNUMP-UHFFFAOYSA-N | $3.6\times10^2$ | | Wang et al. (2017) | Q | 80, 240 |
| MCM:C65OH | $3.2\times10^3$ | | Wang et al. (2017) | Q | 80, 238 |
| $C_6H_{10}O_3$ | $3.4\times10^4$ | | Wang et al. (2017) | Q | 80, 239 |
| ZWRMEYSYDMGCIE-UHFFFAOYSA-N | $1.4\times10^2$ | | Wang et al. (2017) | Q | 80, 240 |
| MCM:C67CHO | $3.2\times10^3$ | | Wang et al. (2017) | Q | 80, 238 |
| $C_6H_{10}O_3$ | $6.2\times10^3$ | | Wang et al. (2017) | Q | 80, 239 |
| BZFZPOBVGBFEBG-UHFFFAOYSA-N | $8.3\times10^1$ | | Wang et al. (2017) | Q | 80, 240 |
| MCM:C6CO2M2OH | $2.3\times10^5$ | | Wang et al. (2017) | Q | 80, 238 |
| $C_6H_{10}O_4$ | $4.4\times10^6$ | | Wang et al. (2017) | Q | 80, 239 |
| ZFPKEBYORKZJID-UHFFFAOYSA-N | $1.6\times10^2$ | | Wang et al. (2017) | Q | 80, 240 |
| MCM:C6CO2OHOOH | $3.4\times10^8$ | | Wang et al. (2017) | Q | 80, 238 |
| $C_6H_{10}O_5$ | $3.3\times10^6$ | | Wang et al. (2017) | Q | 80, 239 |
| LMEOLDNWXYVMAA-UHFFFAOYSA-N | $1.2\times10^2$ | | Wang et al. (2017) | Q | 80, 240 |
| MCM:C6CO3MOH | $5.4\times10^5$ | | Wang et al. (2017) | Q | 80, 238 |
| $C_6H_8O_4$ | $8.3\times10^4$ | | Wang et al. (2017) | Q | 80, 239 |
| WBLLHOUAAGAGCV-UHFFFAOYSA-N | $4.0$ | | Wang et al. (2017) | Q | 80, 240 |
| MCM:C6COHOCHO | $8.3\times10^4$ | | Wang et al. (2017) | Q | 80, 238 |
| $C_6H_{10}O_3$ | $4.0\times10^5$ | | Wang et al. (2017) | Q | 80, 239 |
| HJNIDOQIVMJXHS-UHFFFAOYSA-N | $2.5\times10^4$ | | Wang et al. (2017) | Q | 80, 240 |





Table A3.6: Ketones (RCOR) (...continued)

| Substance / Formula / (Trivial Name) / [CAS Registry Number] / InChIKey | $H_s^{cp}$ (at $T^{\ominus}$) $\left[\dfrac{\text{mol}}{\text{m}^3\,\text{Pa}}\right]$ | $\dfrac{\mathrm{d}\ln H_s^{cp}}{\mathrm{d}(1/T)}$ [K] | Reference | Type | Note |
|---|---|---|---|---|---|
| MCM:C6DICAROOH | $5.5\times10^8$ | | Wang et al. (2017) | Q | 80, 238 |
| $C_6H_{10}O_5$ | $9.6\times10^6$ | | Wang et al. (2017) | Q | 80, 239 |
| HIILQMKIFXFEHC-UHFFFAOYSA-N | $2.0\times10^4$ | | Wang et al. (2017) | Q | 80, 240 |
| MCM:C6O132OH | $1.2\times10^3$ | | Wang et al. (2017) | Q | 80, 238 |
| $C_6H_{10}O_3$ | $3.6\times10^3$ | | Wang et al. (2017) | Q | 80, 239 |
| GIDOBZXUXDNRDY-UHFFFAOYSA-N | 3.0 | | Wang et al. (2017) | Q | 80, 240 |
| MCM:C6TONOHOOH | $2.6\times10^{11}$ | | Wang et al. (2017) | Q | 80, 238 |
| $C_6H_8O_6$ | $1.1\times10^{10}$ | | Wang et al. (2017) | Q | 80, 239 |
| CIXJOARRJVNUNZ-UHFFFAOYSA-N | $1.8\times10^4$ | | Wang et al. (2017) | Q | 80, 240 |
| MCM:H2M3CO4CHO | $1.1\times10^5$ | | Wang et al. (2017) | Q | 80, 238 |
| $C_6H_{10}O_3$ | $2.6\times10^4$ | | Wang et al. (2017) | Q | 80, 239 |
| MIIVJRGFVJVTGE-UHFFFAOYSA-N | 9.8 | | Wang et al. (2017) | Q | 80, 240 |
| MCM:H3C25C5CHO | $3.4\times10^6$ | | Wang et al. (2017) | Q | 80, 238 |
| $C_6H_8O_4$ | $2.0\times10^6$ | | Wang et al. (2017) | Q | 80, 239 |
| CLPDOVMSDUESLY-UHFFFAOYSA-N | $1.0\times10^3$ | | Wang et al. (2017) | Q | 80, 240 |
| MCM:HM22COCHO | $6.2\times10^4$ | | Wang et al. (2017) | Q | 80, 238 |
| $C_6H_{10}O_3$ | $1.8\times10^4$ | | Wang et al. (2017) | Q | 80, 239 |
| UTYMQWKKMYQTHJ-UHFFFAOYSA-N | 6.0 | | Wang et al. (2017) | Q | 80, 240 |
| MCM:MC52CO2OH | $2.3\times10^5$ | | Wang et al. (2017) | Q | 80, 238 |
| $C_6H_{10}O_4$ | $4.6\times10^6$ | | Wang et al. (2017) | Q | 80, 239 |
| LGFUKTGFFSPNKZ-UHFFFAOYSA-N | $1.6\times10^2$ | | Wang et al. (2017) | Q | 80, 240 |
| MCM:MC5COOHOOH | $3.4\times10^8$ | | Wang et al. (2017) | Q | 80, 238 |
| $C_6H_{10}O_5$ | $6.0\times10^6$ | | Wang et al. (2017) | Q | 80, 239 |
| XFHCIPDXRRXAPB-UHFFFAOYSA-N | $8.3\times10^2$ | | Wang et al. (2017) | Q | 80, 240 |
| MCM:MIBKHO4CHO | $6.2\times10^4$ | | Wang et al. (2017) | Q | 80, 238 |
| $C_6H_{10}O_3$ | $1.3\times10^4$ | | Wang et al. (2017) | Q | 80, 239 |
| OYDFBAKICJVUQJ-UHFFFAOYSA-N | $1.4\times10^1$ | | Wang et al. (2017) | Q | 80, 240 |
| MCM:C61CHO | $6.3\times10^5$ | | Wang et al. (2017) | Q | 80, 238 |
| $C_7H_{10}O_4$ | $1.4\times10^5$ | | Wang et al. (2017) | Q | 80, 239 |
| BDFRSVKDVGXKLG-UHFFFAOYSA-N | $2.3\times10^1$ | | Wang et al. (2017) | Q | 80, 240 |
| MCM:C62CHO | $7.3\times10^5$ | | Wang et al. (2017) | Q | 80, 238 |
| $C_7H_{10}O_4$ | $1.3\times10^5$ | | Wang et al. (2017) | Q | 80, 239 |
| QCTBGQUOJXZEIH-UHFFFAOYSA-N | 4.3 | | Wang et al. (2017) | Q | 80, 240 |
| MCM:C6COOHCO3H | $3.0\times10^9$ | | Wang et al. (2017) | Q | 80, 238 |
| $C_7H_8O_6$ | $2.4\times10^9$ | | Wang et al. (2017) | Q | 80, 239 |
| BNWBZUAKPODUMK-UHFFFAOYSA-N | $3.7\times10^1$ | | Wang et al. (2017) | Q | 80, 240 |
| MCM:C6M5CO2OH | $1.1\times10^4$ | | Wang et al. (2017) | Q | 80, 238 |
| $C_7H_{10}O_3$ | $2.2\times10^4$ | | Wang et al. (2017) | Q | 80, 239 |
| YAAUSHWRBSRFSJ-UHFFFAOYSA-N | $1.2\times10^3$ | | Wang et al. (2017) | Q | 80, 240 |



Table A3.6: Ketones (RCOR) (...continued)

| Substance Formula (Trivial Name) [CAS Registry Number] InChIKey | $H_s^{cp}$ (at $T^{\ominus}$) $\left[\dfrac{\text{mol}}{\text{m}^3\,\text{Pa}}\right]$ | $\dfrac{\text{d}\ln H_s^{cp}}{\text{d}(1/T)}$ [K] | Reference | Type | Note |
|---|---|---|---|---|---|
| MCM:C715CO2OH | $1.7\times10^4$ | | Wang et al. (2017) | Q | 80, 238 |
| $C_7H_{10}O_3$ | $7.6\times10^4$ | | Wang et al. (2017) | Q | 80, 239 |
| KBBGTWJMXYAKIO-UHFFFAOYSA-N | $1.1\times10^3$ | | Wang et al. (2017) | Q | 80, 240 |
| MCM:C716OH | $2.8\times10^6$ | | Wang et al. (2017) | Q | 80, 238 |
| $C_7H_{10}O_4$ | $6.8\times10^6$ | | Wang et al. (2017) | Q | 80, 239 |
| QFGSQPDLKNLDDS-UHFFFAOYSA-N | $9.6\times10^4$ | | Wang et al. (2017) | Q | 80, 240 |
| MCM:C717OH | $5.3\times10^7$ | | Wang et al. (2017) | Q | 80, 238 |
| $C_7H_{10}O_4$ | $1.6\times10^7$ | | Wang et al. (2017) | Q | 80, 239 |
| SEENFYUXRUQNHQ-UHFFFAOYSA-N | $2.1\times10^3$ | | Wang et al. (2017) | Q | 80, 240 |
| MCM:C718OH | $5.5\times10^4$ | | Wang et al. (2017) | Q | 80, 238 |
| $C_7H_{12}O_3$ | $2.7\times10^4$ | | Wang et al. (2017) | Q | 80, 239 |
| ALYGFZDZOCWXMF-UHFFFAOYSA-N | $3.6\times10^2$ | | Wang et al. (2017) | Q | 80, 240 |
| MCM:C731OH | $8.0\times10^4$ | | Wang et al. (2017) | Q | 80, 238 |
| $C_7H_{12}O_3$ | $1.7\times10^6$ | | Wang et al. (2017) | Q | 80, 239 |
| KHEBQEUBDOAJRG-UHFFFAOYSA-N | $8.3\times10^3$ | | Wang et al. (2017) | Q | 80, 240 |
| MCM:C733CO | $5.3\times10^7$ | | Wang et al. (2017) | Q | 80, 238 |
| $C_7H_{10}O_4$ | $1.9\times10^7$ | | Wang et al. (2017) | Q | 80, 239 |
| IRSIMXCKMCQRGW-UHFFFAOYSA-N | $1.8\times10^4$ | | Wang et al. (2017) | Q | 80, 240 |
| MCM:C733OH | $1.2\times10^7$ | | Wang et al. (2017) | Q | 80, 238 |
| $C_7H_{12}O_4$ | $3.3\times10^8$ | | Wang et al. (2017) | Q | 80, 239 |
| WRHSSXYTISGZIA-UHFFFAOYSA-N | $3.7\times10^5$ | | Wang et al. (2017) | Q | 80, 240 |
| MCM:C733OOH | $7.4\times10^9$ | | Wang et al. (2017) | Q | 80, 238 |
| $C_7H_{12}O_5$ | $7.3\times10^8$ | | Wang et al. (2017) | Q | 80, 239 |
| OWEUBRYSVXGVIC-UHFFFAOYSA-N | $2.7\times10^5$ | | Wang et al. (2017) | Q | 80, 240 |
| MCM:C735OH | $2.2\times10^6$ | | Wang et al. (2017) | Q | 80, 238 |
| $C_7H_{10}O_4$ | $1.7\times10^7$ | | Wang et al. (2017) | Q | 80, 239 |
| PHOAHMZZKPNPAM-UHFFFAOYSA-N | $4.6\times10^3$ | | Wang et al. (2017) | Q | 80, 240 |
| MCM:C736OH | $8.3\times10^3$ | | Wang et al. (2017) | Q | 80, 238 |
| $C_7H_{10}O_3$ | $1.3\times10^5$ | | Wang et al. (2017) | Q | 80, 239 |
| QWXLVLIZAXRZBR-UHFFFAOYSA-N | $3.2\times10^3$ | | Wang et al. (2017) | Q | 80, 240 |
| MCM:C7ADCOH | $1.3\times10^5$ | | Wang et al. (2017) | Q | 80, 238 |
| $C_7H_{12}O_4$ | $2.8\times10^6$ | | Wang et al. (2017) | Q | 80, 239 |
| SVQINHOHTZDOCH-UHFFFAOYSA-N | $7.8\times10^1$ | | Wang et al. (2017) | Q | 80, 240 |
| MCM:C7ADCOOH | $1.9\times10^8$ | | Wang et al. (2017) | Q | 80, 238 |
| $C_7H_{12}O_5$ | $5.1\times10^6$ | | Wang et al. (2017) | Q | 80, 239 |
| COQUITHNSRNVBJ-UHFFFAOYSA-N | $1.2\times10^2$ | | Wang et al. (2017) | Q | 80, 240 |
| MCM:C7CO3OHOOH | $3.0\times10^{11}$ | | Wang et al. (2017) | Q | 80, 238 |
| $C_7H_{10}O_6$ | $3.2\times10^8$ | | Wang et al. (2017) | Q | 80, 239 |
| YAEIVOJGWQGWPS-UHFFFAOYSA-N | $1.1\times10^4$ | | Wang et al. (2017) | Q | 80, 240 |





Table A3.6: Ketones (RCOR) (...continued)

| Substance Formula (Trivial Name) [CAS Registry Number] InChIKey | $H_s^{cp}$ (at $T^\ominus$) $\left[\dfrac{\mathrm{mol}}{\mathrm{m}^3\,\mathrm{Pa}}\right]$ | $\dfrac{\mathrm{d}\ln H_s^{cp}}{\mathrm{d}(1/T)}$ [K] | Reference | Type | Note |
|---|---|---|---|---|---|
| MCM:C7DCOH | $2.8\times10^5$ | | Wang et al. (2017) | Q | 80, 238 |
| $C_7H_{12}O_4$ | $1.9\times10^6$ | | Wang et al. (2017) | Q | 80, 239 |
| VIPBNCDHLKDPDM-UHFFFAOYSA-N | $1.2\times10^2$ | | Wang et al. (2017) | Q | 80, 240 |
| MCM:C7DCOOH | $4.4\times10^8$ | | Wang et al. (2017) | Q | 80, 238 |
| $C_7H_{12}O_5$ | $5.1\times10^6$ | | Wang et al. (2017) | Q | 80, 239 |
| AVRZSWZUKXCJRK-UHFFFAOYSA-N | $1.0\times10^3$ | | Wang et al. (2017) | Q | 80, 240 |
| MCM:C7DDCOH | $1.8\times10^5$ | | Wang et al. (2017) | Q | 80, 238 |
| $C_7H_{12}O_4$ | $2.6\times10^6$ | | Wang et al. (2017) | Q | 80, 239 |
| HYCGRSOMYJQICU-UHFFFAOYSA-N | $6.6\times10^1$ | | Wang et al. (2017) | Q | 80, 240 |
| MCM:C7DDCOOH | $3.0\times10^8$ | | Wang et al. (2017) | Q | 80, 238 |
| $C_7H_{12}O_5$ | $1.4\times10^6$ | | Wang et al. (2017) | Q | 80, 239 |
| VSEOFSAKNXJKSZ-UHFFFAOYSA-N | $8.1\times10^1$ | | Wang et al. (2017) | Q | 80, 240 |
| MCM:IC7DCOH | $3.2\times10^5$ | | Wang et al. (2017) | Q | 80, 238 |
| $C_7H_{12}O_4$ | $1.5\times10^6$ | | Wang et al. (2017) | Q | 80, 239 |
| DEDFQTYPJDABTI-UHFFFAOYSA-N | $7.3\times10^1$ | | Wang et al. (2017) | Q | 80, 240 |
| MCM:IC7DCOOH | $5.1\times10^8$ | | Wang et al. (2017) | Q | 80, 238 |
| $C_7H_{12}O_5$ | $5.8\times10^6$ | | Wang et al. (2017) | Q | 80, 239 |
| ZVFWDEWSMKAAIM-UHFFFAOYSA-N | $1.4\times10^4$ | | Wang et al. (2017) | Q | 80, 240 |
| MCM:MC6CO2OH | $1.1\times10^4$ | | Wang et al. (2017) | Q | 80, 238 |
| $C_7H_{10}O_3$ | $1.0\times10^4$ | | Wang et al. (2017) | Q | 80, 239 |
| BNJUZVCUVNAPOM-UHFFFAOYSA-N | $3.3\times10^3$ | | Wang et al. (2017) | Q | 80, 240 |
| MCM:C5M2OHCO3H | $2.1\times10^9$ | | Wang et al. (2017) | Q | 80, 238 |
| $C_8H_{10}O_6$ | $3.2\times10^9$ | | Wang et al. (2017) | Q | 80, 239 |
| BRIBGOXGRALVGI-UHFFFAOYSA-N | $9.6\times10^1$ | | Wang et al. (2017) | Q | 80, 240 |
| MCM:C7M15CO2OH | $1.6\times10^4$ | | Wang et al. (2017) | Q | 80, 238 |
| $C_8H_{12}O_3$ | $3.0\times10^4$ | | Wang et al. (2017) | Q | 80, 239 |
| OEJDBFGEAKTZGI-UHFFFAOYSA-N | $6.3\times10^2$ | | Wang et al. (2017) | Q | 80, 240 |
| MCM:C7M6CO2OH | $9.3\times10^3$ | | Wang et al. (2017) | Q | 80, 238 |
| $C_8H_{12}O_3$ | $1.3\times10^4$ | | Wang et al. (2017) | Q | 80, 239 |
| WXVOVBAGQNBDEE-UHFFFAOYSA-N | $4.1\times10^2$ | | Wang et al. (2017) | Q | 80, 240 |
| MCM:C7MO3OHOOH | $3.0\times10^{11}$ | | Wang et al. (2017) | Q | 80, 238 |
| $C_8H_{12}O_6$ | $2.2\times10^8$ | | Wang et al. (2017) | Q | 80, 239 |
| UOUKTGYLQYQUAO-UHFFFAOYSA-N | $4.3\times10^3$ | | Wang et al. (2017) | Q | 80, 240 |
| MCM:C7OHO2CO3H | $2.8\times10^9$ | | Wang et al. (2017) | Q | 80, 238 |
| $C_8H_{10}O_6$ | $1.7\times10^9$ | | Wang et al. (2017) | Q | 80, 239 |
| AOHPTSIBUCEZIY-UHFFFAOYSA-N | $1.6\times10^1$ | | Wang et al. (2017) | Q | 80, 240 |
| MCM:C815CO2OH | $1.5\times10^4$ | | Wang et al. (2017) | Q | 80, 238 |
| $C_8H_{12}O_3$ | $3.8\times10^4$ | | Wang et al. (2017) | Q | 80, 239 |
| IPCSMQASRIZASM-UHFFFAOYSA-N | $6.2\times10^2$ | | Wang et al. (2017) | Q | 80, 240 |





Table A3.6: Ketones (RCOR) (...continued)

| Substance Formula (Trivial Name) [CAS Registry Number] InChIKey | $H_s^{cp}$ (at $T^{\ominus}$) $\left[\dfrac{\text{mol}}{\text{m}^3\,\text{Pa}}\right]$ | $\dfrac{\text{d}\ln H_s^{cp}}{\text{d}(1/T)}$ [K] | Reference | Type | Note |
|---|---|---|---|---|---|
| MCM:C824CO | $1.5\times10^5$ | | Wang et al. (2017) | Q | 80, 238 |
| $C_8H_{12}O_3$ | $4.6\times10^4$ | | Wang et al. (2017) | Q | 80, 239 |
| BQDQKENHFIFOJZ-UHFFFAOYSA-N | $2.5\times10^2$ | | Wang et al. (2017) | Q | 80, 240 |
| MCM:C83COOHOOH | $2.7\times10^{11}$ | | Wang et al. (2017) | Q | 80, 238 |
| $C_8H_{12}O_6$ | $2.0\times10^8$ | | Wang et al. (2017) | Q | 80, 239 |
| LBBMWNQOJHPRLQ-UHFFFAOYSA-N | $6.6\times10^3$ | | Wang et al. (2017) | Q | 80, 240 |
| MCM:C87OH | $2.4\times10^6$ | | Wang et al. (2017) | Q | 80, 238 |
| $C_8H_{12}O_4$ | $2.1\times10^6$ | | Wang et al. (2017) | Q | 80, 239 |
| DFOLUFIWIMPWQH-UHFFFAOYSA-N | $1.0\times10^3$ | | Wang et al. (2017) | Q | 80, 240 |
| MCM:MC7CO2OH | $9.3\times10^3$ | | Wang et al. (2017) | Q | 80, 238 |
| $C_8H_{12}O_3$ | $6.6\times10^3$ | | Wang et al. (2017) | Q | 80, 239 |
| BSMWBPAKWVVEKL-UHFFFAOYSA-N | $1.9\times10^3$ | | Wang et al. (2017) | Q | 80, 240 |
| MCM:C5MEJCO3H | $1.7\times10^9$ | | Wang et al. (2017) | Q | 80, 238 |
| $C_9H_{12}O_6$ | $2.2\times10^9$ | | Wang et al. (2017) | Q | 80, 239 |
| IZKMPGKTEBWFPK-UHFFFAOYSA-N | $5.8\times10^1$ | | Wang et al. (2017) | Q | 80, 240 |
| MCM:C8O2OHCO3H | $2.6\times10^9$ | | Wang et al. (2017) | Q | 80, 238 |
| $C_9H_{12}O_6$ | $1.3\times10^9$ | | Wang et al. (2017) | Q | 80, 239 |
| MBOWYQZNYXRQRC-UHFFFAOYSA-N | $1.1\times10^1$ | | Wang et al. (2017) | Q | 80, 240 |
| MCM:C8OHO2CO3H | $2.2\times10^9$ | | Wang et al. (2017) | Q | 80, 238 |
| $C_9H_{12}O_6$ | $1.1\times10^9$ | | Wang et al. (2017) | Q | 80, 239 |
| CJLPRBWVJFSMFP-UHFFFAOYSA-N | 8.7 | | Wang et al. (2017) | Q | 80, 240 |
| MCM:C914OH | $5.1\times10^6$ | | Wang et al. (2017) | Q | 80, 238 |
| $C_9H_{12}O_4$ | $8.3\times10^7$ | | Wang et al. (2017) | Q | 80, 239 |
| LFDXAZOJHXEEGS-UHFFFAOYSA-N | $2.8\times10^3$ | | Wang et al. (2017) | Q | 80, 240 |
| MCM:C915OH | $1.5\times10^5$ | | Wang et al. (2017) | Q | 80, 238 |
| $C_9H_{14}O_3$ | $2.8\times10^6$ | | Wang et al. (2017) | Q | 80, 239 |
| FAAQZCMVHKUUSB-UHFFFAOYSA-N | $8.0\times10^4$ | | Wang et al. (2017) | Q | 80, 240 |
| MCM:C916OH | $3.6\times10^7$ | | Wang et al. (2017) | Q | 80, 238 |
| $C_9H_{14}O_4$ | $1.4\times10^7$ | | Wang et al. (2017) | Q | 80, 239 |
| MXDMOPCDPNEDIT-UHFFFAOYSA-N | $6.0\times10^3$ | | Wang et al. (2017) | Q | 80, 240 |
| MCM:C918OH | $1.5\times10^5$ | | Wang et al. (2017) | Q | 80, 238 |
| $C_9H_{14}O_3$ | $4.4\times10^5$ | | Wang et al. (2017) | Q | 80, 239 |
| NFJZVEFTUBPOLY-UHFFFAOYSA-N | $1.3\times10^5$ | | Wang et al. (2017) | Q | 80, 240 |
| MCM:C919OH | $3.6\times10^7$ | | Wang et al. (2017) | Q | 80, 238 |
| $C_9H_{14}O_4$ | $1.5\times10^7$ | | Wang et al. (2017) | Q | 80, 239 |
| ULXJXURNPDAXQL-UHFFFAOYSA-N | $3.8\times10^4$ | | Wang et al. (2017) | Q | 80, 240 |
| MCM:C926OH | $1.4\times10^6$ | | Wang et al. (2017) | Q | 80, 238 |
| $C_9H_{14}O_4$ | $7.3\times10^6$ | | Wang et al. (2017) | Q | 80, 239 |
| DBPGQZSZWKEZHT-UHFFFAOYSA-N | $1.2\times10^5$ | | Wang et al. (2017) | Q | 80, 240 |





Table A3.6: Ketones (RCOR) (...continued)

| Substance Formula (Trivial Name) [CAS Registry Number] InChIKey | $H_s^{cp}$ (at $T^{\ominus}$) $\left[\dfrac{\text{mol}}{\text{m}^3\,\text{Pa}}\right]$ | $\dfrac{\text{d}\ln H_s^{cp}}{\text{d}(1/T)}$ [K] | Reference | Type | Note |
|---|---|---|---|---|---|
| MCM:C930OH | $2.6\times10^7$ | | Wang et al. (2017) | Q | 80, 238 |
| $C_9H_{14}O_4$ | $5.3\times10^6$ | | Wang et al. (2017) | Q | 80, 239 |
| LNIVXHBNNGOOPM-UHFFFAOYSA-N | $3.0\times10^3$ | | Wang et al. (2017) | Q | 80, 240 |
| MCM:LMLKAOH | $2.3\times10^6$ | | Wang et al. (2017) | Q | 80, 238 |
| $C_9H_{14}O_4$ | $2.6\times10^7$ | | Wang et al. (2017) | Q | 80, 239 |
| XDMVMFNHBSPJDJ-UHFFFAOYSA-N | $1.6\times10^5$ | | Wang et al. (2017) | Q | 80, 240 |
| MCM:LMLKBOH | $2.0\times10^6$ | | Wang et al. (2017) | Q | 80, 238 |
| $C_9H_{14}O_4$ | $3.0\times10^7$ | | Wang et al. (2017) | Q | 80, 239 |
| JSAHIYWZESQWRA-UHFFFAOYSA-N | $1.1\times10^6$ | | Wang et al. (2017) | Q | 80, 240 |
| MCM:NORLIMOOH | $3.8\times10^9$ | | Wang et al. (2017) | Q | 80, 238 |
| $C_9H_{16}O_5$ | $3.8\times10^8$ | | Wang et al. (2017) | Q | 80, 239 |
| CUMGKBRSRBORIJ-UHFFFAOYSA-N | $6.6\times10^4$ | | Wang et al. (2017) | Q | 80, 240 |
| MCM:C1010OH | $6.8\times10^4$ | | Wang et al. (2017) | Q | 80, 238 |
| $C_{10}H_{16}O_3$ | $5.3\times10^5$ | | Wang et al. (2017) | Q | 80, 239 |
| NQWGZMQLCPYJCB-UHFFFAOYSA-N | $8.7\times10^3$ | | Wang et al. (2017) | Q | 80, 240 |
| MCM:C1012OH | $3.3\times10^4$ | | Wang et al. (2017) | Q | 80, 238 |
| $C_{10}H_{18}O_3$ | $5.6\times10^5$ | | Wang et al. (2017) | Q | 80, 239 |
| ZSBQFHQMMVQAKS-UHFFFAOYSA-N | $1.6\times10^4$ | | Wang et al. (2017) | Q | 80, 240 |
| MCM:C106OH | $2.5\times10^7$ | | Wang et al. (2017) | Q | 80, 238 |
| $C_{10}H_{16}O_4$ | $8.1\times10^6$ | | Wang et al. (2017) | Q | 80, 239 |
| LWIDJJPVYZTGLI-UHFFFAOYSA-N | $9.3\times10^4$ | | Wang et al. (2017) | Q | 80, 240 |
| MCM:C107OH | $3.7\times10^3$ | | Wang et al. (2017) | Q | 80, 238 |
| $C_{10}H_{16}O_3$ | $5.9\times10^4$ | | Wang et al. (2017) | Q | 80, 239 |
| DUMRJPIWTJFZCT-UHFFFAOYSA-N | $6.2\times10^3$ | | Wang et al. (2017) | Q | 80, 240 |
| MCM:C108OH | $2.5\times10^7$ | | Wang et al. (2017) | Q | 80, 238 |
| $C_{10}H_{16}O_4$ | $3.2\times10^6$ | | Wang et al. (2017) | Q | 80, 239 |
| ZDBVABFEYYIFLR-UHFFFAOYSA-N | $5.5\times10^4$ | | Wang et al. (2017) | Q | 80, 240 |
| MCM:C109OH | $6.0\times10^3$ | | Wang et al. (2017) | Q | 80, 238 |
| $C_{10}H_{16}O_3$ | $4.3\times10^4$ | | Wang et al. (2017) | Q | 80, 239 |
| PUENXTSKKXKPMK-UHFFFAOYSA-N | $2.3\times10^4$ | | Wang et al. (2017) | Q | 80, 240 |
| MCM:LIMALAOH | $3.1\times10^4$ | 15000 | Wieser et al. (2023) | Q | 437 |
| $C_{10}H_{16}O_3$ | $5.8\times10^3$ | | Wang et al. (2017) | Q | 80, 238 |
| VNYSFQVUCPNMDZ-UHFFFAOYSA-N | $3.6\times10^4$ | | Wang et al. (2017) | Q | 80, 239 |
| | $1.0\times10^3$ | | Wang et al. (2017) | Q | 80, 240 |
| MCM:LIMALBOH | $5.6\times10^3$ | | Wang et al. (2017) | Q | 80, 238 |
| $C_{10}H_{16}O_3$ | $1.8\times10^4$ | | Wang et al. (2017) | Q | 80, 239 |
| SMXSHIIAZMJGAE-UHFFFAOYSA-N | $5.6\times10^3$ | | Wang et al. (2017) | Q | 80, 240 |
| MCM:LIMALOH | $2.2\times10^{10}$ | 18000 | Wieser et al. (2023) | Q | 437 |
| $C_{10}H_{18}O_4$ | $4.2\times10^7$ | | Wang et al. (2017) | Q | 80, 238 |
| ISUQULRVPDSRIP-UHFFFAOYSA-N | $2.8\times10^8$ | | Wang et al. (2017) | Q | 80, 239 |
| | $9.3\times10^5$ | | Wang et al. (2017) | Q | 80, 240 |



Table A3.6: Ketones (RCOR) (... continued)

| Substance Formula (Trivial Name) [CAS Registry Number] InChIKey | $H_s^{cp}$ (at $T^{\ominus}$) $\left[\dfrac{\mathrm{mol}}{\mathrm{m}^3\,\mathrm{Pa}}\right]$ | $\dfrac{\mathrm{d}\ln H_s^{cp}}{\mathrm{d}(1/T)}$ [K] | Reference | Type | Note |
|---|---|---|---|---|---|
| MCM:LIMALOOH $C_{10}H_{18}O_5$ OQLBLHLZVAPRGN-UHFFFAOYSA-N | $4.3\times10^{10}$ $3.5\times10^{9}$ $4.0\times10^{8}$ $9.6\times10^{5}$ | 18000 | Wieser et al. (2023) Wang et al. (2017) Wang et al. (2017) Wang et al. (2017) | Q Q Q Q | 437 80, 238 80, 239 80, 240 |
| MCM:PINALOH $C_{10}H_{16}O_3$ NGFXWFWSYUSJMH-UHFFFAOYSA-N | $8.1\times10^{4}$ $6.0\times10^{5}$ $7.1\times10^{3}$ | | Wang et al. (2017) Wang et al. (2017) Wang et al. (2017) | Q Q Q | 80, 238 80, 239 80, 240 |
| MCM:C116OH $C_{11}H_{18}O_3$ JMGXHRKVWURTTA-UHFFFAOYSA-N | $9.8\times10^{4}$ $2.6\times10^{5}$ $5.8\times10^{3}$ | | Wang et al. (2017) Wang et al. (2017) Wang et al. (2017) | Q Q Q | 80, 238 80, 239 80, 240 |
| MCM:C117OH $C_{11}H_{18}O_4$ XYBUENVNXDXQKB-UHFFFAOYSA-N | $8.5\times10^{6}$ $4.9\times10^{8}$ $4.7\times10^{3}$ | | Wang et al. (2017) Wang et al. (2017) Wang et al. (2017) | Q Q Q | 80, 238 80, 239 80, 240 |
| MCM:C117OOH $C_{11}H_{18}O_5$ NUIUIELMTMWFAB-UHFFFAOYSA-N | $5.4\times10^{9}$ $2.0\times10^{8}$ $5.9\times10^{5}$ | | Wang et al. (2017) Wang et al. (2017) Wang et al. (2017) | Q Q Q | 80, 238 80, 239 80, 240 |
| MCM:C118CO $C_{11}H_{16}O_5$ QFBAMJSHDLKQRE-UHFFFAOYSA-N | $1.3\times10^{10}$ $5.6\times10^{8}$ $6.3\times10^{5}$ | | Wang et al. (2017) Wang et al. (2017) Wang et al. (2017) | Q Q Q | 80, 238 80, 239 80, 240 |
| MCM:C118OH $C_{11}H_{18}O_5$ GQNPBTACCNTXDZ-UHFFFAOYSA-N | $6.2\times10^{10}$ $3.6\times10^{10}$ $2.9\times10^{5}$ | | Wang et al. (2017) Wang et al. (2017) Wang et al. (2017) | Q Q Q | 80, 238 80, 239 80, 240 |
| MCM:C118OOH $C_{11}H_{18}O_6$ DZELKNYJHUANPY-UHFFFAOYSA-N | $1.6\times10^{12}$ $3.0\times10^{10}$ $1.4\times10^{5}$ | | Wang et al. (2017) Wang et al. (2017) Wang et al. (2017) | Q Q Q | 80, 238 80, 239 80, 240 |
| MCM:C1210OH $C_{12}H_{20}O_3$ RDCPBXUNBHOYIW-UHFFFAOYSA-N | $7.6\times10^{4}$ $2.6\times10^{6}$ $1.6\times10^{3}$ | | Wang et al. (2017) Wang et al. (2017) Wang et al. (2017) | Q Q Q | 80, 238 80, 239 80, 240 |
| MCM:C1214OH $C_{12}H_{20}O_4$ LBONAAUBRWCFLP-UHFFFAOYSA-N | $7.6\times10^{6}$ $4.7\times10^{8}$ $2.1\times10^{3}$ | | Wang et al. (2017) Wang et al. (2017) Wang et al. (2017) | Q Q Q | 80, 238 80, 239 80, 240 |
| MCM:C1214OOH $C_{12}H_{20}O_5$ RDWLNWMMPUJYPV-UHFFFAOYSA-N | $4.4\times10^{9}$ $1.8\times10^{8}$ $8.3\times10^{4}$ | | Wang et al. (2017) Wang et al. (2017) Wang et al. (2017) | Q Q Q | 80, 238 80, 239 80, 240 |
| MCM:C1215CO $C_{12}H_{18}O_5$ MFAMPSJMWGSBSP-UHFFFAOYSA-N | $1.0\times10^{10}$ $3.2\times10^{9}$ $2.8\times10^{6}$ | | Wang et al. (2017) Wang et al. (2017) Wang et al. (2017) | Q Q Q | 80, 238 80, 239 80, 240 |
| MCM:C1215OH $C_{12}H_{20}O_5$ ICTVPTJMPURCGE-UHFFFAOYSA-N | $4.8\times10^{10}$ $1.1\times10^{11}$ $3.2\times10^{6}$ | | Wang et al. (2017) Wang et al. (2017) Wang et al. (2017) | Q Q Q | 80, 238 80, 239 80, 240 |





Table A3.6: Ketones (RCOR) (. . . continued)

| Substance Formula (Trivial Name) [CAS Registry Number] InChIKey | $H_s^{cp}$ (at $T^{\ominus}$) $\left[\dfrac{\mathrm{mol}}{\mathrm{m^3\,Pa}}\right]$ | $\dfrac{\mathrm{d}\ln H_s^{cp}}{\mathrm{d}(1/T)}$ [K] | Reference | Type | Note |
|---|---|---|---|---|---|
| MCM:C1215OOH | $1.3\times10^{12}$ | | Wang et al. (2017) | Q | 80, 238 |
| $C_{12}H_{20}O_6$ | $4.6\times10^{10}$ | | Wang et al. (2017) | Q | 80, 239 |
| PWIVJRAMQXZVCR-UHFFFAOYSA-N | $5.1\times10^{7}$ | | Wang et al. (2017) | Q | 80, 240 |
| MCM:C128CO | $3.3\times10^{7}$ | | Wang et al. (2017) | Q | 80, 238 |
| $C_{12}H_{18}O_4$ | $8.3\times10^{7}$ | | Wang et al. (2017) | Q | 80, 239 |
| UGSIBTFJEGNWKQ-UHFFFAOYSA-N | $6.6\times10^{4}$ | | Wang et al. (2017) | Q | 80, 240 |
| MCM:C128OH | $1.5\times10^{8}$ | | Wang et al. (2017) | Q | 80, 238 |
| $C_{12}H_{20}O_4$ | $2.8\times10^{9}$ | | Wang et al. (2017) | Q | 80, 239 |
| CFXREDIWINSYMD-UHFFFAOYSA-N | $1.3\times10^{5}$ | | Wang et al. (2017) | Q | 80, 240 |
| MCM:C128OOH | $4.7\times10^{9}$ | | Wang et al. (2017) | Q | 80, 238 |
| $C_{12}H_{20}O_5$ | $2.6\times10^{9}$ | | Wang et al. (2017) | Q | 80, 239 |
| UJLXMZVGYRRHNB-UHFFFAOYSA-N | $3.3\times10^{6}$ | | Wang et al. (2017) | Q | 80, 240 |
| MCM:C129OH | $4.1\times10^{6}$ | | Wang et al. (2017) | Q | 80, 238 |
| $C_{12}H_{18}O_4$ | $3.8\times10^{7}$ | | Wang et al. (2017) | Q | 80, 239 |
| LZLVFWPHHNPPPO-UHFFFAOYSA-N | $1.8\times10^{4}$ | | Wang et al. (2017) | Q | 80, 240 |
| MCM:C1312CO | $2.9\times10^{7}$ | | Wang et al. (2017) | Q | 80, 238 |
| $C_{13}H_{20}O_4$ | $5.9\times10^{7}$ | | Wang et al. (2017) | Q | 80, 239 |
| OXGUTNXNCKSGQJ-UHFFFAOYSA-N | $6.6\times10^{5}$ | | Wang et al. (2017) | Q | 80, 240 |
| MCM:C1312OH | $1.4\times10^{8}$ | | Wang et al. (2017) | Q | 80, 238 |
| $C_{13}H_{22}O_4$ | $1.7\times10^{9}$ | | Wang et al. (2017) | Q | 80, 239 |
| IOOAKPNHTHIGJB-UHFFFAOYSA-N | $2.8\times10^{6}$ | | Wang et al. (2017) | Q | 80, 240 |
| MCM:C1312OOH | $3.6\times10^{9}$ | | Wang et al. (2017) | Q | 80, 238 |
| $C_{13}H_{22}O_5$ | $6.5\times10^{8}$ | | Wang et al. (2017) | Q | 80, 239 |
| IMCVJRKOQRMDFG-UHFFFAOYSA-N | $3.9\times10^{6}$ | | Wang et al. (2017) | Q | 80, 240 |
| MCM:BCLKAOH | $2.1\times10^{6}$ | | Wang et al. (2017) | Q | 80, 238 |
| $C_{14}H_{22}O_4$ | $4.2\times10^{7}$ | | Wang et al. (2017) | Q | 80, 239 |
| OIHLYIMTKSRRQE-UHFFFAOYSA-N | $5.8\times10^{4}$ | | Wang et al. (2017) | Q | 80, 240 |
| MCM:BCLKBOH | $1.9\times10^{6}$ | | Wang et al. (2017) | Q | 80, 238 |
| $C_{14}H_{22}O_4$ | $3.0\times10^{7}$ | | Wang et al. (2017) | Q | 80, 239 |
| LFMWXXSOGPOZAN-UHFFFAOYSA-N | $7.4\times10^{5}$ | | Wang et al. (2017) | Q | 80, 240 |
| MCM:BCLKCOH | $2.1\times10^{6}$ | | Wang et al. (2017) | Q | 80, 238 |
| $C_{14}H_{22}O_4$ | $3.6\times10^{7}$ | | Wang et al. (2017) | Q | 80, 239 |
| WUCWLUFQNHHRAC-UHFFFAOYSA-N | $1.4\times10^{6}$ | | Wang et al. (2017) | Q | 80, 240 |
| MCM:C144OOH | $4.2\times10^{9}$ | | Wang et al. (2017) | Q | 80, 238 |
| $C_{14}H_{24}O_5$ | $6.2\times10^{8}$ | | Wang et al. (2017) | Q | 80, 239 |
| GLUIAMUUMIPXJY-UHFFFAOYSA-N | $5.8\times10^{6}$ | | Wang et al. (2017) | Q | 80, 240 |
| MCM:C146OH | $2.1\times10^{6}$ | | Wang et al. (2017) | Q | 80, 238 |
| $C_{14}H_{22}O_4$ | $3.2\times10^{7}$ | | Wang et al. (2017) | Q | 80, 239 |
| MRXVSIMJFRVVPA-UHFFFAOYSA-N | $2.0\times10^{5}$ | | Wang et al. (2017) | Q | 80, 240 |



Table A3.6: Ketones (RCOR) (...continued)

| Substance Formula (Trivial Name) [CAS Registry Number] InChIKey | $H_s^{cp}$ (at $T^\ominus$) $\left[\dfrac{\mathrm{mol}}{\mathrm{m^3\,Pa}}\right]$ | $\dfrac{\mathrm{d}\ln H_s^{cp}}{\mathrm{d}(1/T)}$ [K] | Reference | Type | Note |
|---|---|---|---|---|---|
| MCM:BCALAOH | $6.3\times10^3$ | | Wang et al. (2017) | Q | 80, 238 |
| $C_{15}H_{24}O_3$ | $3.3\times10^4$ | | Wang et al. (2017) | Q | 80, 239 |
| SOIICZPFVOBBSJ-UHFFFAOYSA-N | $6.3\times10^3$ | | Wang et al. (2017) | Q | 80, 240 |
| MCM:BCALBOH | $5.3\times10^3$ | | Wang et al. (2017) | Q | 80, 238 |
| $C_{15}H_{24}O_3$ | $2.1\times10^4$ | | Wang et al. (2017) | Q | 80, 239 |
| WYNIZHBHTSXQQC-UHFFFAOYSA-N | $5.9\times10^4$ | | Wang et al. (2017) | Q | 80, 240 |
| MCM:BCALCOH | $6.3\times10^3$ | | Wang et al. (2017) | Q | 80, 238 |
| $C_{15}H_{24}O_3$ | $3.4\times10^4$ | | Wang et al. (2017) | Q | 80, 239 |
| NQPQIVDLCLYNNZ-UHFFFAOYSA-N | $1.1\times10^4$ | | Wang et al. (2017) | Q | 80, 240 |
| MCM:BCALOH | $4.4\times10^7$ | | Wang et al. (2017) | Q | 80, 238 |
| $C_{15}H_{26}O_4$ | $4.2\times10^8$ | | Wang et al. (2017) | Q | 80, 239 |
| MPEWUZJEETZOEB-UHFFFAOYSA-N | $5.4\times10^5$ | | Wang et al. (2017) | Q | 80, 240 |
| MCM:BCALOOH | $3.2\times10^9$ | | Wang et al. (2017) | Q | 80, 238 |
| $C_{15}H_{26}O_5$ | $8.7\times10^8$ | | Wang et al. (2017) | Q | 80, 239 |
| HSCKVTFOUFPNNB-UHFFFAOYSA-N | $3.6\times10^7$ | | Wang et al. (2017) | Q | 80, 240 |



### A3.7 Carboxylic acids (RCOOH) and peroxy carboxylic acids (RCOOOH)

Table A3.7: Carboxylic acids (RCOOH) and peroxy carboxylic acids (RCOOOH)

| Substance Formula (Trivial Name) [CAS Registry Number] InChIKey | $H_s^{cp}$ (at $T^{\ominus}$) $\left[\dfrac{\text{mol}}{\text{m}^3\,\text{Pa}}\right]$ | $\dfrac{\text{d}\ln H_s^{cp}}{\text{d}(1/T)}$ [K] | Reference | Type | Note |
|---|---|---|---|---|---|
| methanoic acid | $8.8\times10^1$ | 6100 | Burkholder et al. (2019) | L | |
| HCOOH | $8.8\times10^1$ | 6100 | Burkholder et al. (2015) | L | |
| (formic acid) | $8.8\times10^1$ | 6100 | Sander et al. (2011) | L | |
| [64-18-6] | $8.8\times10^1$ | 6100 | Sander et al. (2006) | L | |
| BDAGIHXWWSANSR-UHFFFAOYSA-N | $6.7\times10^1$ | 5900 | Staudinger and Roberts (2001) | L | |
| | $8.8\times10^1$ | 6100 | Johnson et al. (1996) | M | |
| | $5.4\times10^1$ | | Khan et al. (1995) | M | |
| | $5.4\times10^1$ | 5600 | Khan and Brimblecombe (1992) | M | |
| | $1.3\times10^2$ | | Servant et al. (1991) | M | 487 |
| | $1.5\times10^1$ | | Hwang et al. (1992) | V | |
| | | 5700 | Abraham (1984) | V | |
| | | 5600 | Abraham (1984) | R | 488 |
| | | 5700 | Winiwarter et al. (1988) | T | 489 |
| | $3.7\times10^1$ | 5700 | Jacob (1986) | T | 490 |
| | $5.5\times10^1$ | | Keene and Galloway (1986) | T | |
| | $7.5\times10^1$ | | Johnson (1990) | X | 63 |
| | $5.9\times10^1$ | | Gaffney and Senum (1984) | X | 389, 491 |
| | $5.1\times10^1$ | | Johnson et al. (1996) | C | |
| | $5.1\times10^1$ | | Keene et al. (1995) | C | |
| | $5.3\times10^1$ | | Keene et al. (1995) | C | |
| | $3.7\times10^1$ | 5700 | Lelieveld and Crutzen (1991) | C | |
| | $3.5\times10^1$ | 5700 | Pandis and Seinfeld (1989) | C | |
| | $2.1\times10^2$ | | Keshavarz et al. (2022) | Q | |
| | $4.0\times10^1$ | | Duchowicz et al. (2020) | Q | 184 |
| | $2.2\times10^1$ | | Wang et al. (2017) | Q | 80, 238 |
| | $2.5\times10^2$ | | Wang et al. (2017) | Q | 80, 239 |
| | $1.0\times10^2$ | | Wang et al. (2017) | Q | 80, 240 |
| | $2.3\times10^2$ | | Hilal et al. (2008) | Q | |
| | $1.2\times10^1$ | | Modarresi et al. (2007) | Q | 67 |
| | | 5800 | Kühne et al. (2005) | Q | |
| | $5.8\times10^1$ | | Yaffe et al. (2003) | Q | 248, 249 |
| | $8.6\times10^1$ | | Abraham (2003) | Q | |
| | 7.7 | | Katritzky et al. (1998) | Q | |
| | $5.9\times10^1$ | | Duchowicz et al. (2020) | ? | 185, 21 |
| | | 6500 | Kühne et al. (2005) | ? | |
| | $1.3\times10^1$ | | Yaws (1999) | ? | 21 |
| | 8.9 | | Yaws and Yang (1992) | ? | 21 |
| ethanoic acid | $4.0\times10^1$ | 6200 | Burkholder et al. (2019) | L | |
| CH$_3$COOH | $4.0\times10^1$ | 6200 | Burkholder et al. (2015) | L | |
| (acetic acid) | $4.0\times10^1$ | 6200 | Sander et al. (2011) | L | |
| [64-19-7] | $4.0\times10^1$ | 6200 | Sander et al. (2006) | L | |
| QTBSBXVTEAMEQO-UHFFFAOYSA-N | $4.6\times10^1$ | 6300 | Staudinger and Roberts (2001) | L | |
| | $1.4\times10^1$ | | von Hartungen et al. (2004) | M | |
| | $4.0\times10^1$ | 6300 | Johnson et al. (1996) | M | |



Table A3.7: Carboxylic acids (RCOOH) and peroxy carboxylic acids (RCOOOH) (...continued)

| Substance<br>Formula<br>(Trivial Name)<br>[CAS Registry Number]<br>InChIKey | $H_s^{cp}$<br>(at $T^{\ominus}$)<br>$\left[\dfrac{\text{mol}}{\text{m}^3\,\text{Pa}}\right]$ | $\dfrac{\text{d}\ln H_s^{cp}}{\text{d}(1/T)}$<br><br>[K] | Reference | Type | Note |
|---|---|---|---|---|---|
| | $5.4\times10^1$ | | Khan et al. (1995) | M | |
| | $5.4\times10^1$ | 8300 | Khan and Brimblecombe (1992) | M | |
| | $9.2\times10^1$ | | Servant et al. (1991) | M | 487 |
| | | | Fredenhagen and Liebster (1932) | M | 328 |
| | 9.1 | | Hwang et al. (1992) | V | |
| | | 6300 | Abraham (1984) | V | |
| | | 6200 | Abraham (1984) | R | 488 |
| | $4.8\times10^1$ | 6400 | Plyasunov et al. (2001) | T | |
| | $8.7\times10^1$ | 6400 | Jacob et al. (1989) | T | |
| | | 6400 | Winiwarter et al. (1988) | T | 489 |
| | $8.7\times10^1$ | | Keene and Galloway (1986) | T | |
| | 9.7 | 4900 | Goldstein (1982) | X | 298 |
| | $9.9\times10^1$ | | Gaffney and Senum (1984) | X | 389, 491 |
| | $5.1\times10^1$ | | Johnson et al. (1996) | C | |
| | $5.2\times10^1$ | | Keene et al. (1995) | C | |
| | $8.5\times10^1$ | | Keene et al. (1995) | C | |
| | $1.0\times10^2$ | | Keshavarz et al. (2022) | Q | |
| | $3.1\times10^1$ | | Duchowicz et al. (2020) | Q | |
| | $1.4\times10^1$ | | Wang et al. (2017) | Q | 80, 238 |
| | $2.9\times10^2$ | | Wang et al. (2017) | Q | 80, 239 |
| | $6.5\times10^1$ | | Wang et al. (2017) | Q | 80, 240 |
| | $3.3\times10^1$ | | Li et al. (2014) | Q | 241 |
| | $2.0\times10^1$ | | Raventos-Duran et al. (2010) | Q | 242, 243 |
| | $1.2\times10^2$ | | Raventos-Duran et al. (2010) | Q | 244 |
| | $2.0\times10^1$ | | Raventos-Duran et al. (2010) | Q | 245 |
| | $1.3\times10^2$ | | Hilal et al. (2008) | Q | |
| | $3.2\times10^1$ | | Modarresi et al. (2007) | Q | 67 |
| | | 6100 | Kühne et al. (2005) | Q | |
| | $9.9\times10^1$ | | Yaffe et al. (2003) | Q | 248, 249 |
| | $3.1\times10^1$ | | Abraham (2003) | Q | |
| | $3.1\times10^1$ | | English and Carroll (2001) | Q | 230, 231 |
| | $1.1\times10^1$ | | Katritzky et al. (1998) | Q | |
| | $2.3\times10^1$ | | Russell et al. (1992) | Q | 279 |
| | $3.1\times10^1$ | | Suzuki et al. (1992) | Q | 232 |
| | $3.9\times10^1$ | | Nirmalakhandan and Speece (1988) | Q | |
| | $9.9\times10^1$ | | Duchowicz et al. (2020) | ? | 185, 21 |
| | $1.1\times10^1$ | | Maniere et al. (2011) | ? | 241, 165 |
| | | 6200 | Kühne et al. (2005) | ? | |
| | 8.4 | | Yaws (1999) | ? | 21 |
| | 8.2 | | Yaws and Yang (1992) | ? | 21 |
| | $3.3\times10^1$ | | Abraham et al. (1990) | ? | |
| | $3.3\times10^1$ | | Hine and Mookerjee (1975) | ? | |



Table A3.7: Carboxylic acids (RCOOH) and peroxy carboxylic acids (RCOOOH) (...continued)

| Substance Formula (Trivial Name) [CAS Registry Number] InChIKey | $H_s^{cp}$ (at $T^{\ominus}$) $\left[\dfrac{\text{mol}}{\text{m}^3\,\text{Pa}}\right]$ | $\dfrac{\text{d}\ln H_s^{cp}}{\text{d}(1/T)}$ [K] | Reference | Type | Note |
|---|---|---|---|---|---|
| acetic anhydride | 1.7 | | Duchowicz et al. (2020) | V | 186 |
| C$_4$H$_6$O$_3$ | 3.6 | | Yaws (2003) | X | 237 |
| [108-24-7] | $6.3\times10^{-1}$ | | Duchowicz et al. (2020) | Q | |
| WFDIJRYMOXRFFG-UHFFFAOYSA-N | 1.4 | | Wang et al. (2017) | Q | 80, 238 |
| | $5.3\times10^1$ | | Wang et al. (2017) | Q | 80, 239 |
| | 1.7 | | Wang et al. (2017) | Q | 80, 240 |
| | $7.8\times10^{-1}$ | | Raventos-Duran et al. (2010) | Q | 242, 243 |
| | $3.1\times10^1$ | | Raventos-Duran et al. (2010) | Q | 244 |
| | $3.1\times10^{-1}$ | | Raventos-Duran et al. (2010) | Q | 245 |
| | 3.9 | | Gharagheizi et al. (2010) | Q | 246 |
| | $7.1\times10^{-1}$ | | Modarresi et al. (2007) | Q | 67 |
| | 2.3 | | Yaws (1999) | ? | 21 |
| propanoic acid | $7.1\times10^1$ | | Kim and Kim (2016) | M | |
| C$_2$H$_5$COOH | $1.5\times10^1$ | | von Hartungen et al. (2004) | M | |
| (propionic acid) | $5.6\times10^1$ | | Khan et al. (1995) | M | |
| [79-09-4] | $5.5\times10^1$ | | Khan and Brimblecombe (1992) | M | |
| XBDQKXXYIPTUBI-UHFFFAOYSA-N | $6.1\times10^1$ | | Servant et al. (1991) | M | 487 |
| | $2.2\times10^1$ | | Butler and Ramchandani (1935) | M | |
| | | 6800 | Abraham (1984) | V | |
| | | 6800 | Abraham (1984) | R | 488 |
| | $3.7\times10^1$ | 6800 | Plyasunov et al. (2001) | T | |
| | $2.6\times10^1$ | | Keshavarz et al. (2022) | Q | |
| | $4.2\times10^1$ | | Duchowicz et al. (2020) | Q | 299 |
| | $1.2\times10^1$ | | Wang et al. (2017) | Q | 80, 238 |
| | $2.6\times10^2$ | | Wang et al. (2017) | Q | 80, 239 |
| | $1.6\times10^1$ | | Wang et al. (2017) | Q | 80, 240 |
| | $1.6\times10^1$ | | Raventos-Duran et al. (2010) | Q | 242, 243 |
| | $1.2\times10^2$ | | Raventos-Duran et al. (2010) | Q | 244 |
| | $1.2\times10^1$ | | Raventos-Duran et al. (2010) | Q | 245 |
| | $7.0\times10^1$ | | Hilal et al. (2008) | Q | |
| | $2.8\times10^1$ | | Modarresi et al. (2007) | Q | 67 |
| | $2.4\times10^1$ | | Yaffe et al. (2003) | Q | 248, 249 |
| | $2.2\times10^1$ | | Abraham (2003) | Q | |
| | $2.3\times10^1$ | | English and Carroll (2001) | Q | 230, 231 |
| | 2.4 | | Katritzky et al. (1998) | Q | |
| | 5.6 | | Russell et al. (1992) | Q | 358 |
| | $2.4\times10^1$ | | Suzuki et al. (1992) | Q | 232 |
| | $3.4\times10^1$ | | Nirmalakhandan and Speece (1988) | Q | |
| | $2.2\times10^1$ | | Duchowicz et al. (2020) | ? | 185, 21 |
| | $1.1\times10^1$ | | Yaws (1999) | ? | 21 |
| | $2.2\times10^1$ | | Abraham et al. (1990) | ? | |
| | $2.2\times10^1$ | | Hine and Mookerjee (1975) | ? | |



Table A3.7: Carboxylic acids (RCOOH) and peroxy carboxylic acids (RCOOOH) (...continued)

| Substance Formula (Trivial Name) [CAS Registry Number] InChIKey | $H_s^{cp}$ (at $T^\ominus$) $\left[\dfrac{\text{mol}}{\text{m}^3\,\text{Pa}}\right]$ | $\dfrac{\mathrm{d}\ln H_s^{cp}}{\mathrm{d}(1/T)}$ [K] | Reference | Type | Note |
|---|---|---|---|---|---|
| methoxyacetic acid | $1.5\times10^3$ | | Duchowicz et al. (2020) | V | 186 |
| $C_3H_6O_3$ | $3.0\times10^2$ | | Duchowicz et al. (2020) | Q | |
| [625-45-6] | $3.0\times10^2$ | | Wang et al. (2017) | Q | 80, 238 |
| RMIODHQZRUFFFF-UHFFFAOYSA-N | $6.6\times10^2$ | | Wang et al. (2017) | Q | 80, 239 |
| | $1.0\times10^3$ | | Wang et al. (2017) | Q | 80, 240 |
| butanoic acid | $4.3\times10^1$ | | Kim and Kim (2016) | M | |
| $C_3H_7COOH$ | 9.7 | | von Hartungen et al. (2004) | M | |
| (butyric acid) | $4.7\times10^1$ | | Khan et al. (1995) | M | |
| [107-92-6] | $4.5\times10^1$ | | Khan and Brimblecombe (1992) | M | |
| FERIUCNNQQJTOY-UHFFFAOYSA-N | $1.8\times10^1$ | | Butler and Ramchandani (1935) | M | |
| | 9.4 | | Hwang et al. (1992) | V | |
| | | 7100 | Abraham (1984) | V | |
| | | 7300 | Abraham (1984) | R | 488 |
| | $2.5\times10^1$ | | Keshavarz et al. (2022) | Q | |
| | $5.0\times10^1$ | | Duchowicz et al. (2020) | Q | |
| | 9.1 | | Wang et al. (2017) | Q | 80, 238 |
| | $1.3\times10^2$ | | Wang et al. (2017) | Q | 80, 239 |
| | $1.4\times10^1$ | | Wang et al. (2017) | Q | 80, 240 |
| | $1.2\times10^1$ | | Raventos-Duran et al. (2010) | Q | 271, 243 |
| | $6.2\times10^1$ | | Raventos-Duran et al. (2010) | Q | 244 |
| | 9.9 | | Raventos-Duran et al. (2010) | Q | 245 |
| | $4.4\times10^1$ | | Hilal et al. (2008) | Q | |
| | $2.5\times10^1$ | | Modarresi et al. (2007) | Q | 67 |
| | $1.7\times10^1$ | | Abraham (2003) | Q | |
| | $1.8\times10^1$ | | English and Carroll (2001) | Q | 230, 274 |
| | 2.8 | | Russell et al. (1992) | Q | 279 |
| | $1.8\times10^1$ | | Suzuki et al. (1992) | Q | 232 |
| | $2.7\times10^1$ | | Nirmalakhandan and Speece (1988) | Q | |
| | $1.8\times10^1$ | | Duchowicz et al. (2020) | ? | 185, 21 |
| | $1.8\times10^1$ | | Abraham et al. (1990) | ? | |
| | $1.8\times10^1$ | | Hine and Mookerjee (1975) | ? | |
| 2-methylpropanoic acid | $4.6\times10^1$ | | Kim and Kim (2016) | M | |
| $(CH_3)_2CHCOOH$ | 9.6 | | von Hartungen et al. (2004) | M | |
| (isobutyric acid) | $1.1\times10^1$ | | Khan et al. (1995) | M | |
| [79-31-2] | $1.1\times10^1$ | | Khan and Brimblecombe (1992) | M | |
| KQNPFQTWMSNSAP-UHFFFAOYSA-N | $5.6\times10^1$ | | Servant et al. (1991) | M | 487 |
| | 1.4 | | Mackay et al. (2006c) | V | |
| | $2.5\times10^1$ | | Keshavarz et al. (2022) | Q | |
| | $1.9\times10^1$ | | Duchowicz et al. (2020) | Q | |
| | $1.1\times10^1$ | | Wang et al. (2017) | Q | 80, 238 |
| | $6.3\times10^1$ | | Wang et al. (2017) | Q | 80, 239 |
| | $1.7\times10^1$ | | Wang et al. (2017) | Q | 80, 240 |
| | $2.0\times10^1$ | | Gharagheizi et al. (2012) | Q | |
| | $1.2\times10^1$ | | Raventos-Duran et al. (2010) | Q | 242, 243 |
| | $3.1\times10^1$ | | Raventos-Duran et al. (2010) | Q | 244 |
| | 9.9 | | Raventos-Duran et al. (2010) | Q | 245 |



Table A3.7: Carboxylic acids (RCOOH) and peroxy carboxylic acids (RCOOOH) (...continued)

| Substance Formula (Trivial Name) [CAS Registry Number] InChIKey | $H_s^{cp}$ (at $T^{\ominus}$) $\left[\dfrac{\text{mol}}{\text{m}^3\,\text{Pa}}\right]$ | $\dfrac{\text{d}\ln H_s^{cp}}{\text{d}(1/T)}$ [K] | Reference | Type | Note |
|---|---|---|---|---|---|
| | $2.5\times10^1$ | | Hilal et al. (2008) | Q | |
| | $4.7\times10^1$ | | Modarresi et al. (2007) | Q | 67 |
| | $1.1\times10^1$ | | Duchowicz et al. (2020) | ? | 185, 21 |
| | $1.1\times10^1$ | | Yaws (1999) | ? | 21, 12 |
| pentanoic acid | $2.3\times10^1$ | 6900 | Staudinger and Roberts (2001) | L | |
| C$_4$H$_9$COOH | $2.4\times10^1$ | | Kim and Kim (2016) | M | |
| (valeric acid) | $1.2\times10^1$ | | von Hartungen et al. (2004) | M | |
| [109-52-4] | $2.3\times10^1$ | 6600 | Khan et al. (1995) | M | |
| NQPDZGIKBAWPEJ-UHFFFAOYSA-N | $2.1\times10^1$ | 6900 | Khan and Brimblecombe (1992) | M | |
| | $1.2\times10^1$ | | Mackay et al. (2006c) | V | |
| | $1.2\times10^1$ | | Mackay et al. (1995) | V | |
| | $1.6\times10^1$ | | Brimblecombe et al. (1992) | V | |
| | | 7500 | Abraham (1984) | V | |
| | $1.3\times10^1$ | | Amoore and Buttery (1978) | V | |
| | | 7700 | Abraham (1984) | R | 488 |
| | 7.7 | | Yaws (2003) | X | 237 |
| | $1.2\times10^1$ | | Keshavarz et al. (2022) | Q | |
| | $5.5\times10^1$ | | Duchowicz et al. (2020) | Q | 184 |
| | 8.5 | | Wang et al. (2017) | Q | 80, 238 |
| | $6.6\times10^1$ | | Wang et al. (2017) | Q | 80, 239 |
| | $1.4\times10^1$ | | Wang et al. (2017) | Q | 80, 240 |
| | 9.9 | | Raventos-Duran et al. (2010) | Q | 242, 243 |
| | $3.1\times10^1$ | | Raventos-Duran et al. (2010) | Q | 244 |
| | 7.8 | | Raventos-Duran et al. (2010) | Q | 245 |
| | $1.6\times10^1$ | | Gharagheizi et al. (2010) | Q | 246 |
| | $3.3\times10^1$ | | Hilal et al. (2008) | Q | |
| | $1.8\times10^1$ | | Modarresi et al. (2007) | Q | 67 |
| | | 7200 | Kühne et al. (2005) | Q | |
| | $1.4\times10^1$ | | Yaffe et al. (2003) | Q | 248, 272 |
| | $1.1\times10^1$ | | Abraham (2003) | Q | |
| | $1.4\times10^1$ | | English and Carroll (2001) | Q | 230, 231 |
| | 2.9 | | Katritzky et al. (1998) | Q | |
| | $2.2\times10^1$ | | Nirmalakhandan et al. (1997) | Q | |
| | $2.1\times10^1$ | | Duchowicz et al. (2020) | ? | 185, 21 |
| | | 6900 | Kühne et al. (2005) | ? | |
| | 7.4 | | Yaws (1999) | ? | 21 |
| | $1.3\times10^1$ | | Abraham et al. (1990) | ? | |
| 2-methylbutanoic acid | 6.7 | | Duchowicz et al. (2020) | V | 186 |
| C$_5$H$_{10}$O$_2$ | 4.9 | | Yaws (2003) | X | 237 |
| [116-53-0] | $2.1\times10^1$ | | Duchowicz et al. (2020) | Q | |
| WLAMNBDJUVNPJU-UHFFFAOYSA-N | 8.5 | | Wang et al. (2017) | Q | 80, 238 |
| | $3.3\times10^1$ | | Wang et al. (2017) | Q | 80, 239 |
| | $1.6\times10^1$ | | Wang et al. (2017) | Q | 80, 240 |
| | 8.3 | | Gharagheizi et al. (2010) | Q | 246 |
| | $1.6\times10^1$ | | Hilal et al. (2008) | Q | |
| | $4.4\times10^1$ | | Modarresi et al. (2007) | Q | 67 |





Table A3.7: Carboxylic acids (RCOOH) and peroxy carboxylic acids (RCOOOH) (...continued)

| Substance Formula (Trivial Name) [CAS Registry Number] InChIKey | $H_s^{cp}$ (at $T^\ominus$) $\left[\dfrac{\text{mol}}{\text{m}^3\,\text{Pa}}\right]$ | $\dfrac{\text{d}\ln H_s^{cp}}{\text{d}(1/T)}$ [K] | Reference | Type | Note |
|---|---|---|---|---|---|
| 3-methylbutanoic acid $(CH_3)_2CHCH_2COOH$ (isovaleric acid) [503-74-2] GWYFCOCPABKNJV-UHFFFAOYSA-N | $2.7\times10^1$ | | Kim and Kim (2016) | M | |
| | $1.1\times10^1$ | | von Hartungen et al. (2004) | M | |
| | $1.2\times10^1$ | | Khan et al. (1995) | M | |
| | $1.2\times10^1$ | | Khan and Brimblecombe (1992) | M | |
| | $1.2\times10^1$ | | Amoore and Buttery (1978) | M | |
| | 1.6 | | Mackay et al. (2006c) | V | |
| | 1.6 | | Mackay et al. (1995) | V | |
| | 7.3 | | Amoore and Buttery (1978) | V | |
| | $1.1\times10^1$ | | Yaws (2003) | X | 237, 12 |
| | $1.2\times10^1$ | | Keshavarz et al. (2022) | Q | |
| | $2.1\times10^1$ | | Duchowicz et al. (2020) | Q | 299 |
| | 8.5 | | Wang et al. (2017) | Q | 80, 238 |
| | $6.8\times10^1$ | | Wang et al. (2017) | Q | 80, 239 |
| | $1.7\times10^1$ | | Wang et al. (2017) | Q | 80, 240 |
| | $2.2\times10^1$ | | Gharagheizi et al. (2012) | Q | |
| | 9.9 | | Raventos-Duran et al. (2010) | Q | 242, 243 |
| | $3.9\times10^1$ | | Raventos-Duran et al. (2010) | Q | 244 |
| | 7.8 | | Raventos-Duran et al. (2010) | Q | 245 |
| | 8.4 | | Gharagheizi et al. (2010) | Q | 246 |
| | $2.8\times10^1$ | | Hilal et al. (2008) | Q | |
| | $2.0\times10^1$ | | Modarresi et al. (2007) | Q | 67 |
| | $4.8\times10^1$ | | Yao et al. (2002) | Q | 229 |
| | $1.4\times10^1$ | | English and Carroll (2001) | Q | 230, 231 |
| | $1.2\times10^1$ | | Duchowicz et al. (2020) | ? | 185, 21 |
| | 7.0 | | Yaws (1999) | ? | 21, 12 |
| | $1.2\times10^1$ | | Abraham et al. (1990) | ? | |
| 2,2-dimethylpropanoic acid $(CH_3)_3CCOOH$ (pivalic acid) [75-98-9] IUGYQRQAERSCNH-UHFFFAOYSA-N | 3.5 | | Khan et al. (1995) | M | |
| | 3.5 | | Khan and Brimblecombe (1992) | M | |
| | 8.4 | | Yaws (2003) | X | 237 |
| | $1.2\times10^1$ | | Keshavarz et al. (2022) | Q | |
| | 9.4 | | Duchowicz et al. (2020) | Q | |
| | 9.9 | | Raventos-Duran et al. (2010) | Q | 242, 243 |
| | 9.9 | | Raventos-Duran et al. (2010) | Q | 244 |
| | 7.8 | | Raventos-Duran et al. (2010) | Q | 245 |
| | 6.9 | | Gharagheizi et al. (2010) | Q | 246 |
| | $1.2\times10^1$ | | Hilal et al. (2008) | Q | |
| | $1.6\times10^1$ | | Modarresi et al. (2007) | Q | 67 |
| | 3.8 | | Yaffe et al. (2003) | Q | 248, 249 |
| | $9.0\times10^{-1}$ | | Katritzky et al. (1998) | Q | |
| | 3.6 | | Duchowicz et al. (2020) | ? | 185, 21 |



Table A3.7: Carboxylic acids (RCOOH) and peroxy carboxylic acids (RCOOOH) (...continued)

| Substance Formula (Trivial Name) [CAS Registry Number] InChIKey | $H_s^{cp}$ (at $T^\ominus$) $\left[\dfrac{\mathrm{mol}}{\mathrm{m}^3\,\mathrm{Pa}}\right]$ | $\dfrac{\mathrm{d}\ln H_s^{cp}}{\mathrm{d}(1/T)}$ [K] | Reference | Type | Note |
|---|---|---|---|---|---|
| hexanoic acid | $1.3\times10^1$ | 6100 | Staudinger and Roberts (2001) | L | |
| $C_5H_{11}COOH$ | 9.9 | | Kim and Kim (2016) | M | |
| (caproic acid) | 7.5 | | von Hartungen et al. (2004) | M | |
| [142-62-1] | $1.3\times10^1$ | 6300 | Khan et al. (1995) | M | |
| FUZZWVXGSFPDMH-UHFFFAOYSA-N | $1.3\times10^1$ | 5900 | Khan and Brimblecombe (1992) | M | |
| | $1.7\times10^1$ | | Mackay et al. (2006c) | V | |
| | 1.7 | | Mackay et al. (1995) | V | |
| | $1.1\times10^1$ | | Brimblecombe et al. (1992) | V | |
| | $2.0\times10^1$ | | Hwang et al. (1992) | V | |
| | | 8700 | Abraham (1984) | V | |
| | | 8100 | Abraham (1984) | R | 488 |
| | $2.4\times10^1$ | | Yaws (2003) | X | 237, 12 |
| | 6.1 | | Keshavarz et al. (2022) | Q | |
| | $5.9\times10^1$ | | Duchowicz et al. (2020) | Q | 184 |
| | $1.7\times10^1$ | | Gharagheizi et al. (2012) | Q | |
| | 6.2 | | Raventos-Duran et al. (2010) | Q | 242, 243 |
| | $2.5\times10^1$ | | Raventos-Duran et al. (2010) | Q | 244 |
| | 6.2 | | Raventos-Duran et al. (2010) | Q | 245 |
| | $1.4\times10^1$ | | Gharagheizi et al. (2010) | Q | 246 |
| | $2.4\times10^1$ | | Hilal et al. (2008) | Q | |
| | $1.7\times10^1$ | | Modarresi et al. (2007) | Q | 67 |
| | | 7500 | Kühne et al. (2005) | Q | |
| | $1.4\times10^1$ | | Yaffe et al. (2003) | Q | 248, 249 |
| | 8.3 | | Abraham (2003) | Q | |
| | $3.9\times10^1$ | | Yao et al. (2002) | Q | 229, 267 |
| | $1.1\times10^1$ | | English and Carroll (2001) | Q | 230, 231 |
| | 7.9 | | Katritzky et al. (1998) | Q | |
| | $1.7\times10^1$ | | Nirmalakhandan et al. (1997) | Q | |
| | $1.3\times10^1$ | | Duchowicz et al. (2020) | ? | 185, 21 |
| | | 7200 | Kühne et al. (2005) | ? | |
| | $1.5\times10^1$ | | Yaws (1999) | ? | 21, 12 |
| | $1.5\times10^1$ | | Abraham et al. (1990) | ? | |
| 2-methylpentanoic acid | $1.1\times10^1$ | | Hilal et al. (2008) | Q | |
| $C_6H_{12}O_2$ | $3.6\times10^1$ | | Modarresi et al. (2007) | Q | 67 |
| [97-61-0] | | | | | |
| OVBFMEVBMNZIBR-UHFFFAOYSA-N | | | | | |
| 2-ethylbutanoic acid | 6.2 | | Duchowicz et al. (2020) | V | 186 |
| $C_6H_{12}O_2$ | 4.3 | | Yaws (2003) | X | 237 |
| [88-09-5] | $2.3\times10^1$ | | Duchowicz et al. (2020) | Q | |
| OXQGTIUCKGYOAA-UHFFFAOYSA-N | 8.9 | | Gharagheizi et al. (2012) | Q | |
| | 6.2 | | Raventos-Duran et al. (2010) | Q | 242, 243 |
| | 9.9 | | Raventos-Duran et al. (2010) | Q | 244 |
| | 6.2 | | Raventos-Duran et al. (2010) | Q | 245 |
| | 7.6 | | Gharagheizi et al. (2010) | Q | 246 |
| | 9.0 | | Hilal et al. (2008) | Q | |
| | $3.2\times10^1$ | | Modarresi et al. (2007) | Q | 67 |



Table A3.7: Carboxylic acids (RCOOH) and peroxy carboxylic acids (RCOOOH) (... continued)

| Substance Formula (Trivial Name) [CAS Registry Number] InChIKey | $H_s^{cp}$ (at $T^\ominus$) $\left[\dfrac{\text{mol}}{\text{m}^3\,\text{Pa}}\right]$ | $\dfrac{\text{d}\ln H_s^{cp}}{\text{d}(1/T)}$ [K] | Reference | Type | Note |
|---|---|---|---|---|---|
| | 6.1 | | Yaffe et al. (2003) | Q | 248, 249 |
| | $7.9\times10^{-1}$ | | Katritzky et al. (1998) | Q | |
| heptanoic acid | $1.5\times10^1$ | | Kim and Kim (2016) | M | |
| $C_7H_{14}O_2$ | $1.5\times10^1$ | | Duchowicz et al. (2020) | V | 186 |
| [111-14-8] | 9.6 | | Brimblecombe et al. (1992) | V | |
| MNWFXJYAOYHMED-UHFFFAOYSA-N | | 8500 | Abraham (1984) | V | |
| | | 8500 | Abraham (1984) | R | 488 |
| | $6.1\times10^1$ | | Duchowicz et al. (2020) | Q | |
| | 5.5 | | Wang et al. (2017) | Q | 80, 238 |
| | $2.9\times10^1$ | | Wang et al. (2017) | Q | 80, 239 |
| | $2.0\times10^1$ | | Wang et al. (2017) | Q | 80, 240 |
| | $1.0\times10^1$ | | Gharagheizi et al. (2012) | Q | |
| | $1.7\times10^1$ | | Hilal et al. (2008) | Q | |
| | $1.6\times10^1$ | | Modarresi et al. (2007) | Q | 67 |
| | | 7800 | Kühne et al. (2005) | Q | |
| | $3.0\times10^1$ | | Yaffe et al. (2003) | Q | 248, 249 |
| | 6.4 | | Abraham (2003) | Q | |
| | $2.9\times10^1$ | | Yao et al. (2002) | Q | 229 |
| | 4.3 | | Katritzky et al. (1998) | Q | |
| | | 7900 | Kühne et al. (2005) | ? | |
| | $2.5\times10^1$ | | Yaws (1999) | ? | 21, 12 |
| | $1.3\times10^1$ | | Abraham et al. (1990) | ? | |
| 4,4-dimethylpentanoic acid | 4.3 | | Zhang et al. (2010) | Q | 287, 288 |
| $C_7H_{14}O_2$ | $1.4\times10^1$ | | Zhang et al. (2010) | Q | 287, 289 |
| [95823-36-2] | $1.6\times10^3$ | | Zhang et al. (2010) | Q | 287, 290 |
| HMMSZUQCCUWXRA-UHFFFAOYSA-N | 1.6 | | Zhang et al. (2010) | Q | 287, 291 |
| 2-ethyl-2-methylbutanoic acid | 4.3 | | Zhang et al. (2010) | Q | 287, 288 |
| $C_7H_{14}O_2$ | 5.4 | | Zhang et al. (2010) | Q | 287, 289 |
| [19889-37-3] | $2.3\times10^2$ | | Zhang et al. (2010) | Q | 287, 290 |
| LHJPKLWGGMAUAN-UHFFFAOYSA-N | 1.6 | | Zhang et al. (2010) | Q | 287, 291 |
| octanoic acid | $1.1\times10^1$ | | Duchowicz et al. (2020) | V | 186 |
| $C_8H_{16}O_2$ | $1.5\times10^{-1}$ | | Mackay et al. (2006c) | V | |
| (caprylic acid) | $1.5\times10^{-1}$ | | Mackay et al. (1995) | V | |
| [124-07-2] | 7.6 | | Brimblecombe et al. (1992) | V | |
| WWZKQHOCKIZLMA-UHFFFAOYSA-N | | 9600 | Abraham (1984) | V | |
| | | 8900 | Abraham (1984) | R | 488 |
| | $1.9\times10^1$ | | Yaws (2003) | X | 237, 12 |
| | $6.4\times10^1$ | | Duchowicz et al. (2020) | Q | |
| | 6.5 | | Gharagheizi et al. (2012) | Q | |
| | $1.2\times10^1$ | | Gharagheizi et al. (2010) | Q | 246 |
| | $1.3\times10^1$ | | Hilal et al. (2008) | Q | |
| | $1.2\times10^1$ | | Modarresi et al. (2007) | Q | 67 |
| | | 8200 | Kühne et al. (2005) | Q | |
| | $1.2\times10^1$ | | Yaffe et al. (2003) | Q | 248, 249 |
| | 5.7 | | Abraham (2003) | Q | |



Table A3.7: Carboxylic acids (RCOOH) and peroxy carboxylic acids (RCOOOH) (... continued)

| Substance<br>Formula<br>(Trivial Name)<br>[CAS Registry Number]<br>InChIKey | $H_s^{cp}$ (at $T^\ominus$) $\left[\dfrac{\mathrm{mol}}{\mathrm{m^3\,Pa}}\right]$ | $\dfrac{\mathrm{d}\ln H_s^{cp}}{\mathrm{d}(1/T)}$ [K] | Reference | Type | Note |
|---|---|---|---|---|---|
| | $2.7\times10^1$ | | Yao et al. (2002) | Q | 229 |
| | 5.8 | | Katritzky et al. (1998) | Q | |
| | | 8400 | Kühne et al. (2005) | ? | |
| | $1.0\times10^1$ | | Yaws (1999) | ? | 21, 12 |
| | $1.1\times10^1$ | | Abraham et al. (1990) | ? | |
| 2-ethylhexanoic acid<br>$C_8H_{16}O_2$<br>[149-57-5]<br>OBETXYAYXDNJHR-UHFFFAOYSA-N | 3.5<br>3.4<br>$2.4\times10^1$ | | HSDB (2015)<br>Hilal et al. (2008)<br>Modarresi et al. (2007) | V<br>Q<br>Q | <br><br>67 |
| endothal<br>$C_8H_{10}O_5$<br>[145-73-3]<br>GXEKYRXVRROBEV-UHFFFAOYSA-N | $2.6\times10^{10}$<br>$1.4\times10^6$ | | Duchowicz et al. (2020)<br>Duchowicz et al. (2020) | V<br>Q | 186 |
| nonanoic acid<br>$C_9H_{18}O_2$<br>(pelargic acid)<br>[112-05-0]<br>FBUKVWPVBMHYJY-UHFFFAOYSA-N | 6.1<br>3.8<br>$2.0\times10^1$<br>6.9<br>$6.5\times10^1$<br>4.0<br>$1.3\times10^1$<br>9.9<br>$1.2\times10^1$<br>$1.2\times10^1$<br>4.2<br>$2.3\times10^1$<br>5.6<br>3.0<br>$1.1\times10^1$ | | Duchowicz et al. (2020)<br>Brimblecombe et al. (1992)<br>Yaws (2003)<br>Hilal et al. (2008)<br>Duchowicz et al. (2020)<br>Gharagheizi et al. (2012)<br>Gharagheizi et al. (2010)<br>Hilal et al. (2008)<br>Modarresi et al. (2007)<br>Yaffe et al. (2003)<br>Abraham (2003)<br>Yao et al. (2002)<br>Katritzky et al. (1998)<br>Maniere et al. (2011)<br>Yaws (1999) | V<br>V<br>X<br>C<br>Q<br>Q<br>Q<br>Q<br>Q<br>Q<br>Q<br>Q<br>Q<br>?<br>? | 186<br><br>237, 12<br><br><br><br>246<br><br>67<br>248, 272<br><br>229<br><br>12, 165<br>21, 12 |
| decanoic acid<br>$C_{10}H_{20}O_2$<br>[334-48-5]<br>GHVNFZFCNZKVNT-UHFFFAOYSA-N | 7.4<br>6.5<br>$6.7\times10^1$<br>7.7<br>$1.0\times10^1$<br>3.0<br>$1.9\times10^2$ | | Duchowicz et al. (2020)<br>Hilal et al. (2008)<br>Duchowicz et al. (2020)<br>Hilal et al. (2008)<br>Modarresi et al. (2007)<br>Abraham (2003)<br>Yaws (1999) | V<br>C<br>Q<br>Q<br>Q<br>Q<br>? | 186<br><br><br><br>67<br><br>21, 12 |
| 3,3,5,5-tetramethylhexanoic acid<br>$C_{10}H_{20}O_2$<br>[1135681-77-4]<br>WRPPDRMFGQJMAR-UHFFFAOYSA-N | 1.9<br>3.5<br>$1.0\times10^3$<br>$6.1\times10^{-1}$ | | Zhang et al. (2010)<br>Zhang et al. (2010)<br>Zhang et al. (2010)<br>Zhang et al. (2010) | Q<br>Q<br>Q<br>Q | 287, 288<br>287, 289<br>287, 290<br>287, 291 |
| pinonic acid<br>$C_{10}H_{16}O_3$<br>[473-72-3]<br>SIZDUQQDBXJXLQ-UHFFFAOYSA-N | $6.2\times10^3$<br>$3.0\times10^5$<br>$6.2\times10^5$<br>$5.9\times10^4$ | | Wang et al. (2017)<br>Wang et al. (2017)<br>Wang et al. (2017)<br>Isaacman-VanWertz et al. (2016) | Q<br>Q<br>Q<br>Q | 80, 238<br>80, 239<br>80, 240<br>441 |





Table A3.7: Carboxylic acids (RCOOH) and peroxy carboxylic acids (RCOOOH) (...continued)

| Substance Formula (Trivial Name) [CAS Registry Number] InChIKey | $H_s^{cp}$ (at $T^\ominus$) $\left[\dfrac{\text{mol}}{\text{m}^3\,\text{Pa}}\right]$ | $\dfrac{\mathrm{d}\ln H_s^{cp}}{\mathrm{d}(1/T)}$ [K] | Reference | Type | Note |
|---|---|---|---|---|---|
| undecanoic acid $C_{11}H_{22}O_2$ [112-37-8] ZDPHROOEEOARMN-UHFFFAOYSA-N | 7.7 $1.8\times10^1$ 5.8 2.2 | | Yaws (2003) Gharagheizi et al. (2010) Hilal et al. (2008) Abraham (2003) | X Q Q Q | 237 246 |
| dodecanoic acid $C_{12}H_{24}O_2$ [143-07-7] POULHZVOKOAJMA-UHFFFAOYSA-N | 4.5 1.7 $1.7\times10^1$ $8.4\times10^1$ | | Hilal et al. (2008) Abraham (2003) Yao et al. (2002) Yaws (1999) | Q Q Q ? | 229 21, 12 |
| tridecanoic acid $C_{13}H_{26}O_2$ [638-53-9] SZHOJFHSIKHZHA-UHFFFAOYSA-N | 8.0 $3.8\times10^1$ 1.2 | | Yaws (2003) Gharagheizi et al. (2010) Abraham (2003) | X Q Q | 237 246 |
| tetradecanoic acid $C_{14}H_{28}O_2$ [544-63-8] TUNFSRHWOTWDNC-UHFFFAOYSA-N | $2.0\times10^1$ $7.1\times10^1$ $9.4\times10^{-1}$ | | Duchowicz et al. (2020) Duchowicz et al. (2020) Abraham (2003) | V Q Q | 186 |
| pentadecanoic acid $C_{15}H_{30}O_2$ [1002-84-2] WQEPLUUGTLDZJY-UHFFFAOYSA-N | $7.1\times10^{-1}$ | | Abraham (2003) | Q | |
| hexadecanoic acid $C_{16}H_{32}O_2$ (palmitic acid) [57-10-3] IPCSVZSSVZVIGE-UHFFFAOYSA-N | $4.9\times10^{-1}$ $7.2\times10^1$ 3.0 $5.2\times10^{-1}$ | | Duchowicz et al. (2020) Duchowicz et al. (2020) Gharagheizi et al. (2012) Abraham (2003) | V Q Q Q | 186 |
| heptadecanoic acid $C_{17}H_{34}O_2$ (margaric acid) [506-12-7] KEMQGTRYUADPNZ-UHFFFAOYSA-N | $3.8\times10^{-1}$ | | Abraham (2003) | Q | |
| octadecanoic acid $C_{18}H_{36}O_2$ (stearic acid) [57-11-4] QIQXTHQIDYTFRH-UHFFFAOYSA-N | $2.1\times10^1$ $2.5\times10^5$ $7.3\times10^1$ $8.4\times10^{-1}$ 3.5 $3.0\times10^{-1}$ | | Duchowicz et al. (2020) Mackay et al. (1995) Duchowicz et al. (2020) Hilal et al. (2008) Modarresi et al. (2007) Abraham (2003) | V V Q Q Q Q | 186 67 |
| nonadecanoic acid $C_{19}H_{38}O_2$ [646-30-0] ISYWECDDZWTKFF-UHFFFAOYSA-N | $2.3\times10^{-1}$ | | Abraham (2003) | Q | |



Table A3.7: Carboxylic acids (RCOOH) and peroxy carboxylic acids (RCOOOH) (...continued)

| Substance Formula (Trivial Name) [CAS Registry Number] InChIKey | $H_s^{cp}$ (at $T^{\ominus}$) $\left[\dfrac{\mathrm{mol}}{\mathrm{m}^3\,\mathrm{Pa}}\right]$ | $\dfrac{\mathrm{d}\ln H_s^{cp}}{\mathrm{d}(1/T)}$ [K] | Reference | Type | Note |
|---|---|---|---|---|---|
| eicosanoic acid $C_{20}H_{40}O_2$ (arachidic acid) [506-30-9] VKOBVWXKNCXXDE-UHFFFAOYSA-N | $1.7\times10^{-1}$ | | Abraham (2003) | Q | |
| heneicosanoic acid $C_{21}H_{42}O_2$ [2363-71-5] CKDDRHZIAZRDBW-UHFFFAOYSA-N | $1.3\times10^{-1}$ | | Abraham (2003) | Q | |
| docosanoic acid $C_{22}H_{44}O_2$ (behenic acid) [112-85-6] UKMSUNONTOPOIO-UHFFFAOYSA-N | $9.5\times10^{-2}$ | | Abraham (2003) | Q | |
| tricosanoic acid $C_{23}H_{46}O_2$ [2433-96-7] XEZVDURJDFGERA-UHFFFAOYSA-N | $7.2\times10^{-2}$ | | Abraham (2003) | Q | |
| tetracosanoic acid $C_{24}H_{48}O_2$ (lignoceric acid) [557-59-5] QZZGJDVWLFXDLK-UHFFFAOYSA-N | $5.4\times10^{-2}$ | | Abraham (2003) | Q | |
| propenoic acid $C_3H_4O_2$ (acrylic acid) [79-10-7] NIXOWILDQLNWCW-UHFFFAOYSA-N | $2.7\times10^{1}$ | | Duchowicz et al. (2020) | V | 186 |
| | $3.1\times10^{1}$ | | Lide and Frederikse (1995) | V | |
| | $1.2\times10^{2}$ | | Duchowicz et al. (2020) | Q | |
| | $3.3\times10^{1}$ | | Wang et al. (2017) | Q | 80, 238 |
| | $2.2\times10^{1}$ | | Wang et al. (2017) | Q | 80, 239 |
| | $4.6\times10^{1}$ | | Wang et al. (2017) | Q | 80, 240 |
| | $1.6\times10^{1}$ | | Raventos-Duran et al. (2010) | Q | 242, 243 |
| | $9.9$ | | Raventos-Duran et al. (2010) | Q | 244 |
| | $3.1\times10^{1}$ | | Raventos-Duran et al. (2010) | Q | 245 |
| | $2.2\times10^{1}$ | | Hilal et al. (2008) | Q | |
| | $9.4\times10^{2}$ | | Modarresi et al. (2007) | Q | 67 |
| | $2.4\times10^{1}$ | | Yaws (1999) | ? | 21 |
| | $2.4\times10^{1}$ | | Yaws and Yang (1992) | ? | 21 |
| 2-butenoic acid $C_4H_6O_2$ (crotonic acid) [3724-65-0] LDHQCZJRKDOVOX-UHFFFAOYSA-N | $2.3\times10^{1}$ | | Duchowicz et al. (2020) | V | 186 |
| | $5.1\times10^{1}$ | | Duchowicz et al. (2020) | Q | |
| | $4.2\times10^{1}$ | | Wang et al. (2017) | Q | 80, 238 |
| | $2.2\times10^{1}$ | | Wang et al. (2017) | Q | 80, 239 |
| | $5.0\times10^{1}$ | | Wang et al. (2017) | Q | 80, 240 |
| | $2.3\times10^{1}$ | | Hilal et al. (2008) | Q | |
| | $1.1\times10^{2}$ | | Modarresi et al. (2007) | Q | 67 |





Table A3.7: Carboxylic acids (RCOOH) and peroxy carboxylic acids (RCOOOH) (. . . continued)

| Substance<br>Formula<br>(Trivial Name)<br>[CAS Registry Number]<br>InChIKey | $H_s^{cp}$<br>(at $T^{\ominus}$)<br>$\left[\dfrac{\text{mol}}{\text{m}^3\,\text{Pa}}\right]$ | $\dfrac{\text{d}\ln H_s^{cp}}{\text{d}(1/T)}$<br><br>[K] | Reference | Type | Note |
|---|---|---|---|---|---|
| (E)-2-butenoic acid<br>$C_4H_6O_2$<br>(trans-crotonic acid)<br>[107-93-7]<br>LDHQCZJRKDOVOX-NSCUHMNNSA-N | $4.1\times10^1$ | | HSDB (2015) | V | |
| (Z)-2-butenoic acid<br>$C_4H_6O_2$<br>(isocrotonic acid)<br>[503-64-0]<br>LDHQCZJRKDOVOX-IHWYPQMZSA-N | $1.5\times10^2$<br>$5.1\times10^1$<br>$1.2\times10^1$<br>$1.2\times10^1$<br>$2.0\times10^1$<br>$2.3\times10^1$<br>$1.1\times10^2$ | | Duchowicz et al. (2020)<br>Duchowicz et al. (2020)<br>Raventos-Duran et al. (2010)<br>Raventos-Duran et al. (2010)<br>Raventos-Duran et al. (2010)<br>Hilal et al. (2008)<br>Modarresi et al. (2007) | V<br>Q<br>Q<br>Q<br>Q<br>Q<br>Q | 186<br><br>242, 243<br>244<br>245<br><br>67 |
| 2-methyl-2-propenoic acid<br>$C_4H_6O_2$<br>(methacrylic acid)<br>[79-41-4]<br>CERQOIWHTDAKMF-UHFFFAOYSA-N | $2.5\times10^1$<br>1.0<br>9.2<br>$5.0\times10^1$<br>$2.1\times10^1$<br>$1.7\times10^1$<br>8.3<br>$1.2\times10^1$<br>9.9<br>$2.0\times10^1$<br>$1.9\times10^1$<br>$1.4\times10^2$<br>$2.5\times10^1$ | | Khan et al. (1992)<br>Mackay et al. (2006c)<br>Keshavarz et al. (2022)<br>Duchowicz et al. (2020)<br>Wang et al. (2017)<br>Wang et al. (2017)<br>Wang et al. (2017)<br>Raventos-Duran et al. (2010)<br>Raventos-Duran et al. (2010)<br>Raventos-Duran et al. (2010)<br>Hilal et al. (2008)<br>Modarresi et al. (2007)<br>Duchowicz et al. (2020) | M<br>V<br>Q<br>Q<br>Q<br>Q<br>Q<br>Q<br>Q<br>Q<br>Q<br>Q<br>? | <br><br><br>299<br>80, 238<br>80, 239<br>80, 240<br>242, 243<br>244<br>245<br><br>67<br>185, 21 |
| methacrylic acid epoxide<br>$C_4H_6O_3$<br>CSEUSVYSDPXJAP-UHFFFAOYSA-N | $1.2\times10^3$ | | Pye et al. (2013) | Q | 492 |
| benzenecarboxylic acid<br>$C_6H_5COOH$<br>(benzoic acid)<br>[65-85-0]<br>WPYMKLBDIGXBTP-UHFFFAOYSA-N | $2.9\times10^2$<br>$2.6\times10^2$<br>$2.5\times10^2$<br>$1.4\times10^2$<br>2.1<br>$1.7\times10^2$<br>$1.4\times10^2$<br>$1.4\times10^2$<br>$3.1\times10^2$<br>$4.6\times10^2$<br>$5.1\times10^2$<br>$3.5\times10^2$<br>$1.2\times10^2$<br>$1.6\times10^2$<br>$9.9\times10^1$<br> | <br><br><br><br><br><br>6500<br>6700 | Li et al. (2007)<br>Duchowicz et al. (2020)<br>Mackay et al. (2006c)<br>Lide and Frederikse (1995)<br>Mackay et al. (1995)<br>Meylan and Howard (1991)<br>Goldstein (1982)<br>Howard (1989)<br>Duchowicz et al. (2020)<br>Wang et al. (2017)<br>Wang et al. (2017)<br>Wang et al. (2017)<br>Raventos-Duran et al. (2010)<br>Raventos-Duran et al. (2010)<br>Raventos-Duran et al. (2010)<br>Hilal et al. (2008)<br>Kühne et al. (2005) | M<br>V<br>V<br>V<br>V<br>V<br>X<br>X<br>Q<br>Q<br>Q<br>Q<br>Q<br>Q<br>Q<br>Q<br>Q | <br>186<br><br><br><br><br>298<br>412<br><br>80, 238<br>80, 239<br>80, 240<br>242, 243<br>244<br>245<br><br> |





Table A3.7: Carboxylic acids (RCOOH) and peroxy carboxylic acids (RCOOOH) (. . . continued)

| Substance<br>Formula<br>(Trivial Name)<br>[CAS Registry Number]<br>InChIKey | $H_s^{cp}$<br>(at $T^{\ominus}$)<br>$\left[\dfrac{\text{mol}}{\text{m}^3\,\text{Pa}}\right]$ | $\dfrac{\text{d}\ln H_s^{cp}}{\text{d}(1/T)}$<br>[K] | Reference | Type | Note |
|---|---|---|---|---|---|
| (E)-2-butenoic acid<br>$C_4H_6O_2$<br>(trans-crotonic acid)<br>[107-93-7]<br>LDHQCZJRKDOVOX-NSCUHMNNSA-N | $4.1\times10^1$ | | HSDB (2015) | V | |
| (Z)-2-butenoic acid<br>$C_4H_6O_2$<br>(isocrotonic acid)<br>[503-64-0]<br>LDHQCZJRKDOVOX-IHWYPQMZSA-N | $1.5\times10^2$<br>$5.1\times10^1$<br>$1.2\times10^1$<br>$1.2\times10^1$<br>$2.0\times10^1$<br>$2.3\times10^1$<br>$1.1\times10^2$ | | Duchowicz et al. (2020)<br>Duchowicz et al. (2020)<br>Raventos-Duran et al. (2010)<br>Raventos-Duran et al. (2010)<br>Raventos-Duran et al. (2010)<br>Hilal et al. (2008)<br>Modarresi et al. (2007) | V<br>Q<br>Q<br>Q<br>Q<br>Q<br>Q | 186<br><br>242, 243<br>244<br>245<br><br>67 |
| 2-methyl-2-propenoic acid<br>$C_4H_6O_2$<br>(methacrylic acid)<br>[79-41-4]<br>CERQOIWHTDAKMF-UHFFFAOYSA-N | $2.5\times10^1$<br>1.0<br>9.2<br>$5.0\times10^1$<br>$2.1\times10^1$<br>$1.7\times10^1$<br>8.3<br>$1.2\times10^1$<br>9.9<br>$2.0\times10^1$<br>$1.9\times10^1$<br>$1.4\times10^2$<br>$2.5\times10^1$ | | Khan et al. (1992)<br>Mackay et al. (2006c)<br>Keshavarz et al. (2022)<br>Duchowicz et al. (2020)<br>Wang et al. (2017)<br>Wang et al. (2017)<br>Wang et al. (2017)<br>Raventos-Duran et al. (2010)<br>Raventos-Duran et al. (2010)<br>Raventos-Duran et al. (2010)<br>Hilal et al. (2008)<br>Modarresi et al. (2007)<br>Duchowicz et al. (2020) | M<br>V<br>Q<br>Q<br>Q<br>Q<br>Q<br>Q<br>Q<br>Q<br>Q<br>Q<br>? | <br><br><br>299<br>80, 238<br>80, 239<br>80, 240<br>242, 243<br>244<br>245<br><br>67<br>185, 21 |
| methacrylic acid epoxide<br>$C_4H_6O_3$<br>CSEUSVYSDPXJAP-UHFFFAOYSA-N | $1.2\times10^3$ | | Pye et al. (2013) | Q | 492 |
| benzenecarboxylic acid<br>$C_6H_5COOH$<br>(benzoic acid)<br>[65-85-0]<br>WPYMKLBDIGXBTP-UHFFFAOYSA-N | $2.9\times10^2$<br>$2.6\times10^2$<br>$2.5\times10^2$<br>$1.4\times10^2$<br>2.1<br>$1.7\times10^2$<br>$1.4\times10^2$<br>$1.4\times10^2$<br>$3.1\times10^2$<br>$4.6\times10^2$<br>$5.1\times10^2$<br>$3.5\times10^2$<br>$1.2\times10^2$<br>$1.6\times10^2$<br>$9.9\times10^1$<br> | <br><br><br><br><br><br>6500<br>6500<br><br><br><br><br><br><br><br><br>6700 | Li et al. (2007)<br>Duchowicz et al. (2020)<br>Mackay et al. (2006c)<br>Lide and Frederikse (1995)<br>Mackay et al. (1995)<br>Meylan and Howard (1991)<br>Goldstein (1982)<br>Howard (1989)<br>Duchowicz et al. (2020)<br>Wang et al. (2017)<br>Wang et al. (2017)<br>Wang et al. (2017)<br>Raventos-Duran et al. (2010)<br>Raventos-Duran et al. (2010)<br>Raventos-Duran et al. (2010)<br>Hilal et al. (2008)<br>Kühne et al. (2005) | M<br>V<br>V<br>V<br>V<br>V<br>X<br>X<br>Q<br>Q<br>Q<br>Q<br>Q<br>Q<br>Q<br>Q<br>Q | <br>186<br><br><br><br><br>298<br>412<br><br>80, 238<br>80, 239<br>80, 240<br>242, 243<br>244<br>245<br><br> |



Table A3.7: Carboxylic acids (RCOOH) and peroxy carboxylic acids (RCOOOH) (. . . continued)

| Substance Formula (Trivial Name) [CAS Registry Number] InChIKey | $H_s^{cp}$ (at $T^\ominus$) $\left[\dfrac{\text{mol}}{\text{m}^3\,\text{Pa}}\right]$ | $\dfrac{\text{d}\ln H_s^{cp}}{\text{d}(1/T)}$ [K] | Reference | Type | Note |
|---|---|---|---|---|---|
| | $9.1\times10^1$ | | Meylan and Howard (1991) | Q | |
| | $1.5\times10^3$ | | Maniere et al. (2011) | ? | 12, 165 |
| | | 6200 | Kühne et al. (2005) | ? | |
| | $2.4\times10^2$ | | Yaws and Yang (1992) | ? | 21 |
| sorbic acid $C_6H_8O_2$ [110-44-1] WSWCOQWTEOXDQX-MQQKCMAXSA-N | $3.3\times10^1$ $2.0\times10^2$ | | Abraham et al. (2019) HSDB (2015) | Q Q | 99 |
| $D$(-)-isoascorbic acid $C_6H_8O_6$ (erythorbic acid) [89-65-6] CIWBSHSKHKDKBQ-DUZGATOHSA-N | $2.4\times10^2$ | | HSDB (2015) | Q | 99 |
| shikimic acid $C_7H_{10}O_5$ [138-59-0] JXOHGGNKMLTUBP-UHFFFAOYSA-N | $3.7\times10^8$ | | HSDB (2015) | Q | 99 |
| 4-hydroxybenzoic acid $C_7H_6O_3$ [99-96-7] FJKROLUGYXJWQN-UHFFFAOYSA-N | $1.4\times10^6$ $1.4\times10^6$ $2.7\times10^6$ | | Duchowicz et al. (2020) HSDB (2015) Duchowicz et al. (2020) | V V Q | 186 |
| 3,4,5-trihydroxybenzoic acid $C_7H_6O_5$ (gallic acid) [149-91-7] LNTHITQWFMADLM-UHFFFAOYSA-N | $1.2\times10^{14}$ | | HSDB (2015) | Q | 99 |
| 2-methylbenzoic acid $C_8H_8O_2$ ($o$-toluic acid) [118-90-1] ZWLPBLYKEWSWPD-UHFFFAOYSA-N | 9.7 $2.7\times10^2$ $8.0\times10^1$ $9.3\times10^1$ $8.2\times10^1$ $3.2\times10^1$ $9.9\times10^1$ $1.1\times10^2$ | | Abraham et al. (2019) Wang et al. (2017) Wang et al. (2017) Wang et al. (2017) Zhang et al. (2010) Zhang et al. (2010) Zhang et al. (2010) Zhang et al. (2010) | Q Q Q Q Q Q Q Q | 80, 238 80, 239 80, 240 287, 288 287, 289 287, 290 287, 291 |
| 3-methylbenzoic acid $C_8H_8O_2$ ($m$-toluic acid) [99-04-7] GPSDUZXPYCFOSQ-UHFFFAOYSA-N | 6.6 $1.4\times10^{-1}$ $3.9\times10^1$ $2.7\times10^2$ $3.7\times10^2$ $3.6\times10^2$ $8.2\times10^1$ $1.2\times10^2$ $5.1\times10^2$ $1.1\times10^2$ | | Mackay et al. (2006c) Mackay et al. (1995) Abraham et al. (2019) Wang et al. (2017) Wang et al. (2017) Wang et al. (2017) Zhang et al. (2010) Zhang et al. (2010) Zhang et al. (2010) Zhang et al. (2010) | V V Q Q Q Q Q Q Q Q | 80, 238 80, 239 80, 240 287, 288 287, 289 287, 290 287, 291 |



Table A3.7: Carboxylic acids (RCOOH) and peroxy carboxylic acids (RCOOOH) (...continued)

| Substance Formula (Trivial Name) [CAS Registry Number] InChIKey | $H_s^{cp}$ (at $T^{\ominus}$) $\left[\dfrac{\mathrm{mol}}{\mathrm{m}^3\,\mathrm{Pa}}\right]$ | $\dfrac{\mathrm{d}\ln H_s^{cp}}{\mathrm{d}(1/T)}$ [K] | Reference | Type | Note |
|---|---|---|---|---|---|
| 4-methylbenzoic acid | $2.7\times10^2$ | | Wang et al. (2017) | Q | 80, 238 |
| $C_8H_8O_2$ | $4.2\times10^2$ | | Wang et al. (2017) | Q | 80, 239 |
| (*p*-toluic acid) | $4.7\times10^2$ | | Wang et al. (2017) | Q | 80, 240 |
| [99-94-5] | $8.2\times10^1$ | | Zhang et al. (2010) | Q | 287, 288 |
| LPNBBFKOUUSUDB-UHFFFAOYSA-N | $1.4\times10^2$ | | Zhang et al. (2010) | Q | 287, 289 |
| | $8.8\times10^2$ | | Zhang et al. (2010) | Q | 287, 290 |
| | $1.1\times10^2$ | | Zhang et al. (2010) | Q | 287, 291 |
| | | 7000 | Kühne et al. (2005) | Q | |
| | | 7500 | Kühne et al. (2005) | ? | |
| 2-hydroxy-benzoic acid | $1.3\times10^3$ | | Duchowicz et al. (2020) | V | 186 |
| $C_7H_6O_3$ | $8.0\times10^2$ | | Mackay et al. (2006c) | V | |
| (salicylic acid) | $6.9\times10^2$ | | Mackay et al. (1995) | V | |
| [69-72-7] | 1.8 | | Mackay et al. (1995) | V | |
| YGSDEFSMJLZEOE-UHFFFAOYSA-N | $9.0\times10^1$ | | Yaws (2003) | X | 237 |
| | $1.5\times10^3$ | | Duchowicz et al. (2020) | Q | |
| | $1.6\times10^3$ | | Raventos-Duran et al. (2010) | Q | 242, 243 |
| | $4.9\times10^2$ | | Raventos-Duran et al. (2010) | Q | 244 |
| | $6.2\times10^2$ | | Raventos-Duran et al. (2010) | Q | 245 |
| | $8.9\times10^1$ | | Gharagheizi et al. (2010) | Q | 246 |
| benzeneethanoic acid | $2.4\times10^2$ | | Duchowicz et al. (2020) | V | 186 |
| $C_8H_8O_2$ | $1.5\times10^2$ | | Mackay et al. (2006c) | V | |
| (phenylacetic acid) | $1.8\times10^2$ | | Mackay et al. (1995) | V | |
| [103-82-2] | $1.4\times10^1$ | | Mackay et al. (1995) | V | |
| WLJVXDMOQOGPHL-UHFFFAOYSA-N | $3.4\times10^2$ | | Duchowicz et al. (2020) | Q | |
| | $4.1\times10^2$ | | Wang et al. (2017) | Q | 80, 238 |
| | $2.1\times10^3$ | | Wang et al. (2017) | Q | 80, 239 |
| | $3.6\times10^3$ | | Wang et al. (2017) | Q | 80, 240 |
| | $1.2\times10^3$ | | Raventos-Duran et al. (2010) | Q | 242, 243 |
| | $1.2\times10^3$ | | Raventos-Duran et al. (2010) | Q | 244 |
| | $2.0\times10^2$ | | Raventos-Duran et al. (2010) | Q | 245 |
| | $9.9\times10^2$ | | Hilal et al. (2008) | Q | |
| | $3.2\times10^2$ | | Modarresi et al. (2007) | Q | 67 |
| | $2.4\times10^2$ | | Yaffe et al. (2003) | Q | 248, 249 |
| | $5.1\times10^1$ | | Katritzky et al. (1998) | Q | |
| phthalic anhydride | $6.1\times10^2$ | | Duchowicz et al. (2020) | V | 186 |
| $C_8H_4O_3$ | $1.6\times10^3$ | | Lide and Frederikse (1995) | V | |
| [85-44-9] | $1.2\times10^1$ | | Yaws (2003) | X | 237 |
| LGRFSURHDFAFJT-UHFFFAOYSA-N | $4.4\times10^1$ | | Duchowicz et al. (2020) | Q | |
| | $1.2\times10^1$ | | Gharagheizi et al. (2010) | Q | 246 |
| 1,2-benzenedicarboxylic acid | $4.9\times10^5$ | | Duchowicz et al. (2020) | V | 186 |
| $C_8H_6O_4$ | $4.9\times10^5$ | | HSDB (2015) | V | |
| (phthalic acid) | $8.2\times10^6$ | | Duchowicz et al. (2020) | Q | |
| [88-99-3] | | | | | |
| XNGIFLGASWRNHJ-UHFFFAOYSA-N | | | | | |



Table A3.7: Carboxylic acids (RCOOH) and peroxy carboxylic acids (RCOOOH) (…continued)

| Substance<br>Formula<br>(Trivial Name)<br>[CAS Registry Number]<br>InChIKey | $H_s^{cp}$ (at $T^\ominus$) $\left[\dfrac{\text{mol}}{\text{m}^3\,\text{Pa}}\right]$ | $\dfrac{\text{d}\ln H_s^{cp}}{\text{d}(1/T)}$ [K] | Reference | Type | Note |
|---|---|---|---|---|---|
| terephthalic acid<br>$C_8H_6O_4$<br>[100-21-0]<br>KKEYFWRCBNTPAC-UHFFFAOYSA-N | $2.5\times10^7$<br>$4.8\times10^9$ | | HSDB (2015)<br>Gharagheizi et al. (2012) | Q<br>Q | 447 |
| isophthalic acid<br>$C_8H_6O_4$<br>[121-91-5]<br>QQVIHTHCMHWDBS-UHFFFAOYSA-N | $4.0\times10^9$<br>$7.3\times10^4$<br>$4.5\times10^6$<br>$7.1\times10^8$<br>$3.7\times10^9$ | | Yaws (2003)<br>Abraham et al. (2019)<br>HSDB (2015)<br>Gharagheizi et al. (2012)<br>Gharagheizi et al. (2010) | X<br>Q<br>Q<br>Q<br>Q | 237<br><br>99<br><br>246 |
| dehydroacetic acid<br>$C_8H_8O_4$<br>[520-45-6]<br>PGRHXDWITVMQBC-UHFFFAOYSA-N | $2.9\times10^1$ | | HSDB (2015) | V | |
| 2-methoxybenzoic acid<br>$C_8H_8O_3$<br>[579-75-9]<br>ILUJQPXNXACGAN-UHFFFAOYSA-N | $2.5\times10^3$ | | Abraham et al. (2019) | Q | |
| 4-methoxybenzoic acid<br>$C_8H_8O_3$<br>[100-09-4]<br>ZEYHEAKUIGZSGI-UHFFFAOYSA-N | $1.8\times10^3$ | | Abraham et al. (2019) | Q | |
| caffeic acid<br>$C_9H_8O_4$<br>[331-39-5]<br>QAIPRVGONGVQAS-DUXPYHPUSA-N | $7.0\times10^{10}$ | | HSDB (2015) | Q | 99 |
| 4-methylphthalic anhydride<br>$C_9H_6O_3$<br>[19438-61-0]<br>ZOXBWJMCXHTKNU-UHFFFAOYSA-N | $1.4$<br>$6.4\times10^4$<br>$3.5\times10^1$<br>$3.6\times10^1$ | | Zhang et al. (2010)<br>Zhang et al. (2010)<br>Zhang et al. (2010)<br>Zhang et al. (2010) | Q<br>Q<br>Q<br>Q | 287, 288<br>287, 289<br>287, 290<br>287, 291 |
| 3-phenyl-2-propenoic acid<br>$C_9H_8O_2$<br>(cinnamic acid)<br>[621-82-9]<br>WBYWAXJHAXSJNI-UHFFFAOYSA-N | $5.8\times10^2$<br>$4.1\times10^2$<br>$9.9\times10^2$<br>$1.6\times10^2$<br>$7.8\times10^2$ | | Duchowicz et al. (2020)<br>Duchowicz et al. (2020)<br>Raventos-Duran et al. (2010)<br>Raventos-Duran et al. (2010)<br>Raventos-Duran et al. (2010) | V<br>Q<br>Q<br>Q<br>Q | 186<br><br>242, 243<br>244<br>245 |
| *trans*-cinnamic acid<br>$C_9H_8O_2$<br>[140-10-3]<br>WBYWAXJHAXSJNI-VOTSOKGWSA-N | $2.8\times10^{11}$ | | Abraham et al. (2019) | Q | |
| *p*-coumaric acid<br>$C_9H_8O_3$<br>[501-98-4]<br>NGSWKAQJJWESNS-ZZXKWVIFSA-N | $7.2\times10^5$ | | Abraham et al. (2019) | Q | |





Table A3.7: Carboxylic acids (RCOOH) and peroxy carboxylic acids (RCOOOH) (. . . continued)

| Substance<br>Formula<br>(Trivial Name)<br>[CAS Registry Number]<br>InChIKey | $H_s^{cp}$<br>(at $T^{\ominus}$)<br>$\left[\dfrac{\text{mol}}{\text{m}^3\,\text{Pa}}\right]$ | $\dfrac{\text{d}\ln H_s^{cp}}{\text{d}(1/T)}$<br><br>[K] | Reference | Type | Note |
|---|---|---|---|---|---|
| 3,4-dimethoxybenzoic acid<br>$C_9H_{10}O_4$<br>(veratric acid)<br>[93-07-2]<br>DAUAQNGYDSHRET-UHFFFAOYSA-N | $1.1\times10^5$ | | Abraham et al. (2019) | Q | |
| 3,4,5-trimethoxybenzoic acid<br>$C_{10}H_{12}O_5$<br>(eudesmic acid)<br>[118-41-2]<br>SJSOFNCYXJUNBT-UHFFFAOYSA-N | $7.3\times10^5$ | | Abraham et al. (2019) | Q | |
| 1-naphthaleneacetic acid<br><br>$C_{12}H_{10}O_2$<br><br>[86-87-3]<br>PRPINYUDVPFIRX-UHFFFAOYSA-N | $3.3\times10^2$<br><br>$7.9\times10^3$ | | Maniere et al. (2011)<br><br>Maniere et al. (2011) | ?<br><br>? | 12, 493,<br>165<br>12, 494,<br>165 |
| 2-naphthoxyacetic acid<br>$C_{12}H_{10}O_3$<br>[120-23-0]<br>RZCJYMOBWVJQGV-UHFFFAOYSA-N | $3.5\times10^5$ | | Ebert et al. (2023) | ? | 318 |
| p-tert-butylbenzoic acid<br>$C_{11}H_{14}O_2$<br>[98-73-7]<br>KDVYCTOWXSLNNI-UHFFFAOYSA-N | $3.5\times10^1$<br>$4.5\times10^1$<br>$3.6\times10^2$<br>$4.2\times10^1$ | | Zhang et al. (2010)<br>Zhang et al. (2010)<br>Zhang et al. (2010)<br>Zhang et al. (2010) | Q<br>Q<br>Q<br>Q | 287, 288<br>287, 289<br>287, 290<br>287, 291 |
| trinexapac<br>$C_{11}H_{12}O_5$<br>[104273-73-6]<br>DFFWZNDCNBOKDI-UHFFFAOYSA-N | $3.9\times10^6$<br>$4.3\times10^8$ | | Duchowicz et al. (2020)<br>Duchowicz et al. (2020) | V<br>Q | 186 |
| benzoic acid, anhydride<br>$C_{14}H_{10}O_3$<br>[93-97-0]<br>CHIHQLCVLOXUJW-UHFFFAOYSA-N | $7.0$<br>$3.7\times10^2$<br>$6.5\times10^3$<br>$6.4\times10^2$ | | Zhang et al. (2010)<br>Zhang et al. (2010)<br>Zhang et al. (2010)<br>Zhang et al. (2010) | Q<br>Q<br>Q<br>Q | 287, 288<br>287, 289<br>287, 290<br>287, 291 |
| pyromellitic acid<br>$C_{10}H_6O_8$<br>[89-05-4]<br>CYIDZMCFTVVTJO-UHFFFAOYSA-N | $1.6\times10^{13}$<br>$3.0\times10^{13}$<br>$1.6\times10^{13}$ | | Yaws (2003)<br>Gharagheizi et al. (2012)<br>Gharagheizi et al. (2010) | X<br>Q<br>Q | 237, 495<br><br>246 |
| pyromellitic dianhydride<br>$C_{10}H_2O_6$<br>[89-32-7]<br>ANSXAPJVJOKRDJ-UHFFFAOYSA-N | $1.3\times10^3$<br>$1.3\times10^3$<br>$1.4\times10^{11}$<br>$4.8\times10^4$<br>$9.7\times10^5$ | | HSDB (2015)<br>Zhang et al. (2010)<br>Zhang et al. (2010)<br>Zhang et al. (2010)<br>Zhang et al. (2010) | Q<br>Q<br>Q<br>Q<br>Q | 99<br>287, 288<br>287, 289<br>287, 290<br>287, 291 |





Table A3.7: Carboxylic acids (RCOOH) and peroxy carboxylic acids (RCOOOH) (. . . continued)

| Substance Formula (Trivial Name) [CAS Registry Number] InChIKey | $H_s^{cp}$ (at $T^{\ominus}$) $\left[\dfrac{\text{mol}}{\text{m}^3\,\text{Pa}}\right]$ | $\dfrac{\text{d}\ln H_s^{cp}}{\text{d}(1/T)}$ [K] | Reference | Type | Note |
|---|---|---|---|---|---|
| naproxen C$_{14}$H$_{14}$O$_3$ [22204-53-1] CMWTZPSULFXXJA-VIFPVBQESA-N | $2.5\times10^5$ | | Abraham et al. (2019) | Q | |
| ($Z,Z$)-9,12-octadecadienoic acid C$_{18}$H$_{32}$O$_2$ (linoleic acid) [60-33-3] OYHQOLUKZRVURQ-HZJYTTRNSA-N | $4.9\times10^1$ | | HSDB (2015) | V | |
| rosmarinic acid C$_{18}$H$_{16}$O$_8$ [537-15-5] DOUMFZQKYFQNTF-WUTVXBCWSA-N | $3.7\times10^{21}$ | | HSDB (2015) | Q | 447 |
| abietic acid C$_{20}$H$_{30}$O$_2$ [514-10-3] RSWGJHLUYNHPMX-ONCXSQPRSA-N | $1.5\times10^5$ $1.7\times10^5$ | | Yaws (2003) Gharagheizi et al. (2010) | X Q | 237, 12 246 |
| ethanedioic acid HOOCCOOH (oxalic acid) [144-62-7] MUBZPKHOEPUJKR-UHFFFAOYSA-N | $6.1\times10^6$ $6.1\times10^6$ $6.1\times10^6$ $7.1\times10^6$ $3.1\times10^4$ $4.2\times10^5$ $6.9\times10^4$ $1.6\times10^4$ $1.6\times10^3$ $3.7\times10^7$ $4.9\times10^4$ $2.0\times10^4$ $3.9\times10^5$ $4.2\times10^5$ $2.4\times10^3$ $2.3\times10^5$ $4.1\times10^5$ $4.9\times10^6$ $6.9\times10^4$ | 9800 7300 | Burkholder et al. (2019) Burkholder et al. (2015) Compernolle and Müller (2014a) Clegg et al. (1996) Brimblecombe et al. (1992) Yaws (2003) Gaffney and Senum (1984) Keshavarz et al. (2022) Duchowicz et al. (2020) Gharagheizi et al. (2012) Raventos-Duran et al. (2010) Raventos-Duran et al. (2010) Raventos-Duran et al. (2010) Gharagheizi et al. (2010) Hilal et al. (2008) Modarresi et al. (2007) Meylan and Howard (1991) Saxena and Hildemann (1996) Duchowicz et al. (2020) | L L V V V X X Q Q Q Q Q Q Q Q Q Q E ? | 237, 12 389, 491 242, 243 244 245 246 67 401 185, 21 |
| propanedioic acid HOOCCH$_2$COOH (malonic acid) [141-82-2] OFOBLEOULBTSOW-UHFFFAOYSA-N | $2.4\times10^8$ $2.4\times10^8$ $3.7\times10^4$ $3.8\times10^8$ $9.3\times10^7$ $2.1\times10^1$ $2.5\times10^5$ $6.2\times10^6$ $2.5\times10^6$ | 11000 14000 | Burkholder et al. (2019) Burkholder et al. (2015) Duchowicz et al. (2020) Compernolle and Müller (2014a) Compernolle and Müller (2014a) Duchowicz et al. (2020) Raventos-Duran et al. (2010) Raventos-Duran et al. (2010) Raventos-Duran et al. (2010) | L L V V V Q Q Q Q | 186 242, 243 244 245 |



Table A3.7: Carboxylic acids (RCOOH) and peroxy carboxylic acids (RCOOOH) (...continued)

| Substance Formula (Trivial Name) [CAS Registry Number] InChIKey | $H_s^{cp}$ (at $T^{\ominus}$) $\left[\dfrac{\text{mol}}{\text{m}^3\,\text{Pa}}\right]$ | $\dfrac{\text{d}\ln H_s^{cp}}{\text{d}(1/T)}$ [K] | Reference | Type | Note |
|---|---|---|---|---|---|
| | $1.2\times10^5$ | | Modarresi et al. (2007) | Q | 67 |
| | $3.9\times10^6$ | | Saxena and Hildemann (1996) | E | 401 |
| butanedioic acid | $3.1\times10^7$ | | Burkholder et al. (2019) | L | |
| HOOC(CH$_2$)$_2$COOH | $3.1\times10^7$ | | Burkholder et al. (2015) | L | |
| (succinic acid) | $2.8\times10^7$ | | Duchowicz et al. (2020) | V | 186 |
| [110-15-6] | $2.7\times10^7$ | | HSDB (2015) | V | |
| KDYFGRWQOYBRFD-UHFFFAOYSA-N | $4.1\times10^7$ | 11000 | Compernolle and Müller (2014a) | V | |
| | $2.0\times10^7$ | 12000 | Compernolle and Müller (2014a) | V | |
| | $7.2\times10^5$ | | Yaws (2003) | X | 237 |
| | $6.5\times10^4$ | | Duchowicz et al. (2020) | Q | |
| | $1.4\times10^7$ | | Gharagheizi et al. (2012) | Q | |
| | $4.9\times10^6$ | | Raventos-Duran et al. (2010) | Q | 242, 243 |
| | $2.5\times10^6$ | | Raventos-Duran et al. (2010) | Q | 244 |
| | $2.0\times10^6$ | | Raventos-Duran et al. (2010) | Q | 245 |
| | $3.2\times10^5$ | | Gharagheizi et al. (2010) | Q | 246 |
| | $3.0\times10^6$ | | Saxena and Hildemann (1996) | E | 401 |
| pentanedioic acid | $3.8\times10^7$ | | Burkholder et al. (2019) | L | |
| HOOC(CH$_2$)$_3$COOH | $3.8\times10^7$ | | Burkholder et al. (2015) | L | |
| (glutaric acid) | $1.9\times10^7$ | | Mentel et al. (2004) | M | |
| [110-94-1] | $5.1\times10^7$ | 12000 | Compernolle and Müller (2014a) | V | |
| JFCQEDHGNNZCLN-UHFFFAOYSA-N | $2.4\times10^7$ | 13000 | Compernolle and Müller (2014a) | V | |
| | $1.0\times10^5$ | | Yaws (2003) | X | 237, 496 |
| | $5.0\times10^6$ | | Gharagheizi et al. (2012) | Q | |
| | $4.9\times10^6$ | | Raventos-Duran et al. (2010) | Q | 271, 243 |
| | $3.9\times10^7$ | | Raventos-Duran et al. (2010) | Q | 244 |
| | $1.2\times10^6$ | | Raventos-Duran et al. (2010) | Q | 245 |
| | $3.5\times10^5$ | | Gharagheizi et al. (2010) | Q | 246 |
| | $2.2\times10^7$ | | Hilal et al. (2008) | Q | |
| | $2.3\times10^5$ | | Modarresi et al. (2007) | Q | 67 |
| | $2.0\times10^6$ | | Saxena and Hildemann (1996) | E | 401 |
| 3-hydroxy glutaric acid C$_5$H$_8$O$_5$ [638-18-6] ZQHYXNSQOIDNTL-UHFFFAOYSA-N | $6.7\times10^8$ | | Isaacman-VanWertz et al. (2016) | Q | 441 |
| hexanedioic acid | $6.6\times10^7$ | | Burkholder et al. (2019) | L | |
| HOOC(CH$_2$)$_4$COOH | $6.6\times10^7$ | | Burkholder et al. (2015) | L | |
| (adipic acid) | $2.1\times10^6$ | | Duchowicz et al. (2020) | V | 186 |
| [124-04-9] | $2.1\times10^6$ | | HSDB (2015) | V | |
| WNLRTRBMVRJNCN-UHFFFAOYSA-N | $6.6\times10^7$ | 13000 | Compernolle and Müller (2014a) | V | |
| | $1.1\times10^1$ | | Lide and Frederikse (1995) | V | |
| | $1.8\times10^5$ | 11000 | Goldstein (1982) | X | 298 |
| | $3.3\times10^6$ | | Duchowicz et al. (2020) | Q | |
| | $2.2\times10^6$ | | Gharagheizi et al. (2012) | Q | |
| | $3.9\times10^6$ | | Raventos-Duran et al. (2010) | Q | 242, 243 |
| | $4.9\times10^7$ | | Raventos-Duran et al. (2010) | Q | 244 |



Table A3.7: Carboxylic acids (RCOOH) and peroxy carboxylic acids (RCOOOH) (. . . continued)

| Substance Formula (Trivial Name) [CAS Registry Number] InChIKey | $H_s^{cp}$ (at $T^{\ominus}$) $\left[\dfrac{\text{mol}}{\text{m}^3\,\text{Pa}}\right]$ | $\dfrac{\text{d}\ln H_s^{cp}}{\text{d}(1/T)}$ [K] | Reference | Type | Note |
|---|---|---|---|---|---|
| | $9.9\times10^5$ | | Raventos-Duran et al. (2010) | Q | 245 |
| | $2.5\times10^7$ | | Hilal et al. (2008) | Q | |
| | $2.5\times10^5$ | | Modarresi et al. (2007) | Q | 67 |
| | $2.0\times10^6$ | | Saxena and Hildemann (1996) | E | 401 |
| tricarballylic acid C$_6$H$_8$O$_6$ [99-14-9] KQTIIICEAUMSDG-UHFFFAOYSA-N | $1.9\times10^9$ | | Isaacman-VanWertz et al. (2016) | Q | 441 |
| heptanedioic acid C$_7$H$_{12}$O$_4$ (pimelic acid) [111-16-0] WLJVNTCWHIRURA-UHFFFAOYSA-N | $2.6\times10^7$ $8.1\times10^7$ $4.1\times10^6$ | 15000 | Duchowicz et al. (2020) Compernolle and Müller (2014a) Duchowicz et al. (2020) | V V Q | 186 |
| 3-acetyl pentanedioic acid C$_7$H$_{10}$O$_5$ [149474-71-5] TZPGYCKKEMNHRS-UHFFFAOYSA-N | $4.5\times10^9$ | | Isaacman-VanWertz et al. (2016) | Q | 441 |
| octanedioic acid C$_8$H$_{14}$O$_4$ (suberic acid) [505-48-6] TYFQFVWCELRYAO-UHFFFAOYSA-N | $1.8\times10^6$ $7.7\times10^7$ $4.8\times10^6$ | 14000 | Duchowicz et al. (2020) Compernolle and Müller (2014a) Duchowicz et al. (2020) | V V Q | 186 |
| 3-methyl-1,2,3-butanetricarboxylic acid C$_8$H$_{12}$O$_6$ [77370-41-3] VMWJGTKDJFMTFZ-UHFFFAOYSA-N | $1.9\times10^8$ | | Isaacman-VanWertz et al. (2016) | Q | 441 |
| nonanedioic acid C$_9$H$_{16}$O$_4$ (azelaic acid) [123-99-9] BDJRBEYXGGNYIS-UHFFFAOYSA-N | $8.9\times10^7$ $3.6\times10^6$ $1.9\times10^5$ $9.3\times10^5$ | 17000 | Compernolle and Müller (2014a) Yaws (2003) Gharagheizi et al. (2012) Gharagheizi et al. (2010) | V X Q Q | 237, 12 246 |
| decanedioic acid C$_{10}$H$_{18}$O$_4$ (sebacic acid) [111-20-6] CXMXRPHRNRROMY-UHFFFAOYSA-N | $7.6\times10^7$ $1.7\times10^6$ $1.6\times10^5$ $1.4\times10^6$ | | Compernolle and Müller (2014a) Yaws (2003) Gharagheizi et al. (2012) Gharagheizi et al. (2010) | V X Q Q | 237, 12 246 |
| dodecanedioic acid C$_{12}$H$_{22}$O$_4$ [693-23-2] TVIDDXQYHWJXFK-UHFFFAOYSA-N | $9.9\times10^5$ | | Ebert et al. (2023) | ? | 316 |



Table A3.7: Carboxylic acids (RCOOH) and peroxy carboxylic acids (RCOOOH) (...continued)

| Substance Formula (Trivial Name) [CAS Registry Number] InChIKey | $H_s^{cp}$ (at $T^\ominus$) $\left[\dfrac{\text{mol}}{\text{m}^3\,\text{Pa}}\right]$ | $\dfrac{\text{d}\ln H_s^{cp}}{\text{d}(1/T)}$ [K] | Reference | Type | Note |
|---|---|---|---|---|---|
| *cis*-butenedioic acid | $1.4\times10^8$ | | Lide and Frederikse (1995) | V | |
| HOOC(CH)$_2$COOH | $1.3\times10^6$ | | Yaws (2003) | X | 237 |
| (maleic acid) | $6.8\times10^6$ | | Gharagheizi et al. (2012) | Q | |
| [110-16-7] | $1.3\times10^6$ | | Gharagheizi et al. (2010) | Q | 246 |
| VZCYOOQTPOCHFL-UPHRSURJSA-N | $9.9\times10^6$ | | Saxena and Hildemann (1996) | E | 401 |
| citraconic acid | $3.6\times10^7$ | | Yaws (2003) | X | 237 |
| C$_5$H$_6$O$_4$ | $4.0\times10^6$ | | Gharagheizi et al. (2012) | Q | |
| [498-23-7] | $3.7\times10^7$ | | Gharagheizi et al. (2010) | Q | 246 |
| HNEGQIOMVPPMNR-IHWYPQMZSA-N | | | | | |
| itaconic acid | $1.3\times10^6$ | | Yaws (2003) | X | 237, 12 |
| C$_5$H$_6$O$_4$ | $1.3\times10^7$ | | Gharagheizi et al. (2012) | Q | |
| [97-65-4] | $1.3\times10^6$ | | Gharagheizi et al. (2010) | Q | 246 |
| LVHBHZANLOWSRM-UHFFFAOYSA-N | | | | | |
| 1,4-cyclohexanedicarboxylic acid | $1.1\times10^6$ | | Yaws (2003) | X | 237, 154 |
| C$_8$H$_{12}$O$_4$ | $2.3\times10^6$ | | Gharagheizi et al. (2012) | Q | |
| [619-82-9] | $1.1\times10^6$ | | Gharagheizi et al. (2010) | Q | 246 |
| PXGZQGDTEZPERC-UHFFFAOYSA-N | | | | | |
| pinic acid | $1.4\times10^6$ | | Wang et al. (2017) | Q | 80, 238 |
| C$_9$H$_{14}$O$_4$ | $7.1\times10^7$ | | Wang et al. (2017) | Q | 80, 239 |
| [473-73-4] | $3.2\times10^7$ | | Wang et al. (2017) | Q | 80, 240 |
| LEVONNIFUFSRKZ-UHFFFAOYSA-N | $1.0\times10^7$ | | Isaacman-VanWertz et al. (2016) | Q | 441 |
| methanoic peroxyacid | $2.9\times10^1$ | | Sauer (1997) | M | 449 |
| HCOOOH | 5.2 | | HSDB (2015) | Q | 99 |
| (peroxyformic acid) | | | | | |
| [107-32-4] | | | | | |
| SCKXCAADGDQQCS-UHFFFAOYSA-N | | | | | |
| ethanoic peroxyacid | 8.3 | 5300 | Burkholder et al. (2019) | L | |
| CH$_3$COOOH | 8.3 | 5300 | Burkholder et al. (2015) | L | |
| (peroxyacetic acid) | 8.3 | 5300 | Sander et al. (2011) | L | |
| [79-21-0] | 7.3 | 5600 | Staudinger and Roberts (2001) | L | |
| KFSLWBXXFJQRDL-UHFFFAOYSA-N | $2.4\times10^1$ | | Sauer (1997) | M | 449 |
| | 8.3 | 5300 | O'Sullivan et al. (1996) | M | |
| | 6.5 | 5900 | Lind and Kok (1994) | M | 52 |
| | $1.1\times10^2$ | | Wang et al. (2017) | Q | 80, 238 |
| | 9.6 | | Wang et al. (2017) | Q | 80, 239 |
| | $1.3\times10^{-1}$ | | Wang et al. (2017) | Q | 80, 240 |
| | 7.8 | | Raventos-Duran et al. (2010) | Q | 242, 243 |
| | 2.5 | | Raventos-Duran et al. (2010) | Q | 244 |
| | 7.8 | | Raventos-Duran et al. (2010) | Q | 245 |
| | $1.8\times10^1$ | | Hilal et al. (2008) | Q | |
| | $9.3\times10^1$ | | Modarresi et al. (2007) | Q | 67 |
| | | 6100 | Kühne et al. (2005) | Q | |
| | | 5300 | Kühne et al. (2005) | ? | |





Table A3.7: Carboxylic acids (RCOOH) and peroxy carboxylic acids (RCOOOH) (...continued)

| Substance<br>Formula<br>(Trivial Name)<br>[CAS Registry Number]<br>InChIKey | $H_s^{cp}$<br>(at $T^\ominus$)<br>$\left[\dfrac{\mathrm{mol}}{\mathrm{m^3\,Pa}}\right]$ | $\dfrac{\mathrm{d}\ln H_s^{cp}}{\mathrm{d}(1/T)}$<br><br>[K] | Reference | Type | Note |
|---|---|---|---|---|---|
| MCM:M2C43CO2H | 8.0 | | Wang et al. (2017) | Q | 80, 238 |
| $C_6H_{12}O_2$ | $1.7\times10^1$ | | Wang et al. (2017) | Q | 80, 239 |
| XFOASZQZPWEJAA-UHFFFAOYSA-N | $1.6\times10^1$ | | Wang et al. (2017) | Q | 80, 240 |
| MCM:M33C3CO2H | 4.9 | | Wang et al. (2017) | Q | 80, 238 |
| $C_6H_{12}O_2$ | $1.7\times10^1$ | | Wang et al. (2017) | Q | 80, 239 |
| VUAXHMVRKOTJKP-UHFFFAOYSA-N | 9.6 | | Wang et al. (2017) | Q | 80, 240 |
| MCM:C721OOH | $1.2\times10^6$ | | Wang et al. (2017) | Q | 80, 238 |
| $C_7H_{12}O_4$ | $2.8\times10^6$ | | Wang et al. (2017) | Q | 80, 239 |
| ZLURETBBUKLNQX-UHFFFAOYSA-N | $6.2\times10^5$ | | Wang et al. (2017) | Q | 80, 240 |
| MCM:C721CO3H | $1.4\times10^7$ | | Wang et al. (2017) | Q | 80, 238 |
| $C_8H_{12}O_5$ | $5.5\times10^6$ | | Wang et al. (2017) | Q | 80, 239 |
| HYEGRGIGVZWWFM-UHFFFAOYSA-N | $1.0\times10^4$ | | Wang et al. (2017) | Q | 80, 240 |
| MCM:C811OOH | $9.8\times10^5$ | | Wang et al. (2017) | Q | 80, 238 |
| $C_8H_{14}O_4$ | $3.3\times10^6$ | | Wang et al. (2017) | Q | 80, 239 |
| RYDSGVCXHLRTGT-UHFFFAOYSA-N | $2.5\times10^6$ | | Wang et al. (2017) | Q | 80, 240 |
| MCM:C823OOH | $8.7\times10^5$ | | Wang et al. (2017) | Q | 80, 238 |
| $C_8H_{14}O_4$ | $3.0\times10^6$ | | Wang et al. (2017) | Q | 80, 239 |
| KIIRSRLFRXTUMU-UHFFFAOYSA-N | $2.9\times10^4$ | | Wang et al. (2017) | Q | 80, 240 |
| MCM:NORPINIC | $1.7\times10^6$ | | Wang et al. (2017) | Q | 80, 238 |
| $C_8H_{12}O_4$ | $2.0\times10^7$ | | Wang et al. (2017) | Q | 80, 239 |
| KLGKVMMWRDYKJM-UHFFFAOYSA-N | $6.3\times10^6$ | | Wang et al. (2017) | Q | 80, 240 |
| MCM:C811CO3H | $1.1\times10^7$ | | Wang et al. (2017) | Q | 80, 238 |
| $C_9H_{14}O_5$ | $3.7\times10^6$ | | Wang et al. (2017) | Q | 80, 239 |
| ZVOGBTSHONFRIW-UHFFFAOYSA-N | $2.9\times10^4$ | | Wang et al. (2017) | Q | 80, 240 |
| MCM:C823CO3H | $1.0\times10^7$ | | Wang et al. (2017) | Q | 80, 238 |
| $C_9H_{14}O_5$ | $1.6\times10^6$ | | Wang et al. (2017) | Q | 80, 239 |
| IESQFENJEQJSTC-UHFFFAOYSA-N | $3.2\times10^4$ | | Wang et al. (2017) | Q | 80, 240 |
| MCM:LIMONIC | $1.2\times10^6$ | | Wang et al. (2017) | Q | 80, 238 |
| $C_9H_{14}O_4$ | $6.6\times10^7$ | | Wang et al. (2017) | Q | 80, 239 |
| JUCWCJYNDRRZKR-UHFFFAOYSA-N | $1.8\times10^7$ | | Wang et al. (2017) | Q | 80, 240 |
| MCM:C137OOH | $8.5\times10^5$ | | Wang et al. (2017) | Q | 80, 238 |
| $C_{13}H_{22}O_4$ | $3.7\times10^6$ | | Wang et al. (2017) | Q | 80, 239 |
| RDYPCLKIKFUKHY-UHFFFAOYSA-N | $4.3\times10^5$ | | Wang et al. (2017) | Q | 80, 240 |
| MCM:C137CO2H | $1.2\times10^6$ | | Wang et al. (2017) | Q | 80, 238 |
| $C_{14}H_{22}O_4$ | $1.7\times10^8$ | | Wang et al. (2017) | Q | 80, 239 |
| IYTMEDMFNUDFPT-UHFFFAOYSA-N | $1.7\times10^8$ | | Wang et al. (2017) | Q | 80, 240 |
| MCM:C137CO3H | $9.8\times10^6$ | | Wang et al. (2017) | Q | 80, 238 |
| $C_{14}H_{22}O_5$ | $2.4\times10^6$ | | Wang et al. (2017) | Q | 80, 239 |
| WIBVBPZGNYNCKK-UHFFFAOYSA-N | $1.8\times10^5$ | | Wang et al. (2017) | Q | 80, 240 |



Table A3.7: Carboxylic acids (RCOOH) and peroxy carboxylic acids (RCOOOH) (...continued)

| Substance Formula (Trivial Name) [CAS Registry Number] InChIKey | $H_s^{cp}$ (at $T^\ominus$) $\left[\dfrac{\text{mol}}{\text{m}^3\,\text{Pa}}\right]$ | $\dfrac{\text{d}\ln H_s^{cp}}{\text{d}(1/T)}$ [K] | Reference | Type | Note |
|---|---|---|---|---|---|
| MCM:TM123BCO2H | $1.6\times10^2$ | | Wang et al. (2017) | Q | 80, 238 |
| $C_9H_{10}O_2$ | $9.1\times10^1$ | | Wang et al. (2017) | Q | 80, 239 |
| RIZUCYSQUWMQLX-UHFFFAOYSA-N | $2.3\times10^2$ | | Wang et al. (2017) | Q | 80, 240 |
| MCM:TM124BCO2H | $1.6\times10^2$ | | Wang et al. (2017) | Q | 80, 238 |
| $C_9H_{10}O_2$ | $4.1\times10^2$ | | Wang et al. (2017) | Q | 80, 239 |
| OPVAJFQBSDUNQA-UHFFFAOYSA-N | $5.6\times10^2$ | | Wang et al. (2017) | Q | 80, 240 |
| MCM:TMBCO2H | $1.6\times10^2$ | | Wang et al. (2017) | Q | 80, 238 |
| $C_9H_{10}O_2$ | $2.6\times10^2$ | | Wang et al. (2017) | Q | 80, 239 |
| UMVOQQDNEYOJOK-UHFFFAOYSA-N | $3.6\times10^2$ | | Wang et al. (2017) | Q | 80, 240 |
| MCM:C2OHOCO2H | $2.1\times10^6$ | | Wang et al. (2017) | Q | 80, 238 |
| $C_3H_6O_4$ | $2.0\times10^7$ | | Wang et al. (2017) | Q | 80, 239 |
| RBNPOMFGQQGHHO-UHFFFAOYSA-N | $2.1\times10^6$ | | Wang et al. (2017) | Q | 80, 240 |
| MCM:HOC2H4CO2H | $3.2\times10^4$ | | Wang et al. (2017) | Q | 80, 238 |
| $C_3H_6O_3$ | $1.1\times10^5$ | | Wang et al. (2017) | Q | 80, 239 |
| ALRHLSYJTWAHJZ-UHFFFAOYSA-N | $7.6\times10^4$ | | Wang et al. (2017) | Q | 80, 240 |
| MCM:HC3CO2H | $1.0\times10^5$ | | Wang et al. (2017) | Q | 80, 238 |
| $C_4H_6O_3$ | $3.6\times10^5$ | | Wang et al. (2017) | Q | 80, 239 |
| RMQJECWPWQIIPW-UHFFFAOYSA-N | $1.7\times10^6$ | | Wang et al. (2017) | Q | 80, 240 |
| MCM:HMACO2H | $6.2\times10^4$ | | Wang et al. (2017) | Q | 80, 238 |
| $C_4H_6O_3$ | $8.3\times10^2$ | | Wang et al. (2017) | Q | 80, 239 |
| AAMTXHVZOHPPQR-UHFFFAOYSA-N | $3.6\times10^4$ | | Wang et al. (2017) | Q | 80, 240 |
| MCM:HO2C3CO2H | $3.0\times10^4$ | | Wang et al. (2017) | Q | 80, 238 |
| $C_4H_8O_3$ | $6.9\times10^4$ | | Wang et al. (2017) | Q | 80, 239 |
| WHBMMWSBFZVSSR-UHFFFAOYSA-N | $6.6\times10^4$ | | Wang et al. (2017) | Q | 80, 240 |
| MCM:HOC3H6CO2H | $2.6\times10^4$ | | Wang et al. (2017) | Q | 80, 238 |
| $C_4H_8O_3$ | $1.4\times10^5$ | | Wang et al. (2017) | Q | 80, 239 |
| SJZRECIVHVDYJC-UHFFFAOYSA-N | $1.0\times10^6$ | | Wang et al. (2017) | Q | 80, 240 |
| MCM:HOIPRCO2H | $3.0\times10^4$ | | Wang et al. (2017) | Q | 80, 238 |
| $C_4H_8O_3$ | $4.5\times10^4$ | | Wang et al. (2017) | Q | 80, 239 |
| DBXBTMSZEOQQDU-UHFFFAOYSA-N | $6.9\times10^4$ | | Wang et al. (2017) | Q | 80, 240 |
| MCM:IPRHOCO2H | $9.6\times10^2$ | | Wang et al. (2017) | Q | 80, 238 |
| $C_4H_8O_3$ | $2.3\times10^4$ | | Wang et al. (2017) | Q | 80, 239 |
| BWLBGMIXKSTLSX-UHFFFAOYSA-N | $5.3\times10^3$ | | Wang et al. (2017) | Q | 80, 240 |
| MCM:C46CO2H | $4.9\times10^4$ | | Wang et al. (2017) | Q | 80, 238 |
| $C_5H_8O_3$ | $1.5\times10^5$ | | Wang et al. (2017) | Q | 80, 239 |
| NJMYQRVWBCSLEU-UHFFFAOYSA-N | $5.0\times10^4$ | | Wang et al. (2017) | Q | 80, 240 |
| MCM:HC4ACO2H | $6.9\times10^4$ | | Wang et al. (2017) | Q | 80, 238 |
| $C_5H_8O_3$ | $2.3\times10^5$ | | Wang et al. (2017) | Q | 80, 239 |
| BERUOTKXCOOJJM-UHFFFAOYSA-N | $6.0\times10^5$ | | Wang et al. (2017) | Q | 80, 240 |



Table A3.7: Carboxylic acids (RCOOH) and peroxy carboxylic acids (RCOOOH) (...continued)

| Substance Formula (Trivial Name) [CAS Registry Number] InChIKey | $H_s^{cp}$ (at $T^{\ominus}$) $\left[\dfrac{\text{mol}}{\text{m}^3\,\text{Pa}}\right]$ | $\dfrac{\text{d}\ln H_s^{cp}}{\text{d}(1/T)}$ [K] | Reference | Type | Note |
|---|---|---|---|---|---|
| MCM:HC4CCO2H $C_5H_8O_3$ NCQCQZXQBYAHBZ-UHFFFAOYSA-N | $6.9\times10^4$ $2.0\times10^5$ $6.5\times10^5$ | | Wang et al. (2017) Wang et al. (2017) Wang et al. (2017) | Q Q Q | 80, 238 80, 239 80, 240 |
| MCM:HM22CO2H $C_5H_{10}O_3$ RDFQSFOGKVZWKF-UHFFFAOYSA-N | $1.7\times10^4$ $2.1\times10^4$ $1.8\times10^4$ | | Wang et al. (2017) Wang et al. (2017) Wang et al. (2017) | Q Q Q | 80, 238 80, 239 80, 240 |
| MCM:HO2C43CO2H $C_5H_{10}O_3$ VEXDRERIMPLZLU-UHFFFAOYSA-N | $2.8\times10^4$ $1.6\times10^4$ $3.2\times10^4$ | | Wang et al. (2017) Wang et al. (2017) Wang et al. (2017) | Q Q Q | 80, 238 80, 239 80, 240 |
| MCM:HO2C4CO2H $C_5H_{10}O_3$ FMHKPLXYWVCLME-UHFFFAOYSA-N | $2.3\times10^4$ $8.0\times10^4$ $8.1\times10^5$ | | Wang et al. (2017) Wang et al. (2017) Wang et al. (2017) | Q Q Q | 80, 238 80, 239 80, 240 |
| MCM:HOBUT2CO2H $C_5H_{10}O_3$ JYTYEGKJKIXWOJ-UHFFFAOYSA-N | $2.3\times10^4$ $3.5\times10^4$ $7.1\times10^5$ | | Wang et al. (2017) Wang et al. (2017) Wang et al. (2017) | Q Q Q | 80, 238 80, 239 80, 240 |
| MCM:C518CO2H $C_6H_{10}O_3$ COPVWTGWHHSRKU-UHFFFAOYSA-N | $4.6\times10^4$ $2.8\times10^4$ $2.0\times10^4$ | | Wang et al. (2017) Wang et al. (2017) Wang et al. (2017) | Q Q Q | 80, 238 80, 239 80, 240 |
| MCM:H2M2C4CO2H $C_6H_{12}O_3$ PQJUMPXLDAZULJ-UHFFFAOYSA-N | $1.4\times10^4$ $3.0\times10^4$ $3.2\times10^5$ | | Wang et al. (2017) Wang et al. (2017) Wang et al. (2017) | Q Q Q | 80, 238 80, 239 80, 240 |
| MCM:H2M3C4CO2H $C_6H_{12}O_3$ BNTHHYUWQILHSA-UHFFFAOYSA-N | $2.2\times10^4$ $2.9\times10^4$ $2.4\times10^5$ | | Wang et al. (2017) Wang et al. (2017) Wang et al. (2017) | Q Q Q | 80, 238 80, 239 80, 240 |
| MCM:HM22C3CO2H $C_6H_{12}O_3$ WLKUATLKPJGHKX-UHFFFAOYSA-N | $1.4\times10^4$ $2.3\times10^4$ $1.6\times10^5$ | | Wang et al. (2017) Wang et al. (2017) Wang et al. (2017) | Q Q Q | 80, 238 80, 239 80, 240 |
| MCM:HM2C43CO2H $C_6H_{12}O_3$ QSMSJHRQUBQMKD-UHFFFAOYSA-N | $2.2\times10^4$ $2.3\times10^4$ $4.7\times10^4$ | | Wang et al. (2017) Wang et al. (2017) Wang et al. (2017) | Q Q Q | 80, 238 80, 239 80, 240 |
| MCM:HM33C3CO2H $C_6H_{12}O_3$ NGRPJOQCQYDOJX-UHFFFAOYSA-N | $1.4\times10^4$ $1.7\times10^4$ $6.6\times10^5$ | | Wang et al. (2017) Wang et al. (2017) Wang et al. (2017) | Q Q Q | 80, 238 80, 239 80, 240 |
| MCM:HO2C54CO2H $C_6H_{12}O_3$ IOXPMKQXROCYFL-UHFFFAOYSA-N | $2.2\times10^4$ $1.6\times10^4$ $6.3\times10^5$ | | Wang et al. (2017) Wang et al. (2017) Wang et al. (2017) | Q Q Q | 80, 238 80, 239 80, 240 |
| MCM:HO3C5CO2H $C_6H_{12}O_3$ ABIKNKURIGPIRJ-UHFFFAOYSA-N | $2.0\times10^4$ $3.7\times10^4$ $6.2\times10^5$ | | Wang et al. (2017) Wang et al. (2017) Wang et al. (2017) | Q Q Q | 80, 238 80, 239 80, 240 |



Table A3.7: Carboxylic acids (RCOOH) and peroxy carboxylic acids (RCOOOH) (...continued)

| Substance<br>Formula<br>(Trivial Name)<br>[CAS Registry Number]<br>InChIKey | $H_s^{cp}$<br>(at $T^{\ominus}$)<br>$\left[\dfrac{\mathrm{mol}}{\mathrm{m^3\,Pa}}\right]$ | $\dfrac{\mathrm{d}\ln H_s^{cp}}{\mathrm{d}(1/T)}$<br><br>[K] | Reference | Type | Note |
|---|---|---|---|---|---|
| MCM:C622CO2H | $3.6\times10^4$ | | Wang et al. (2017) | Q | 80, 238 |
| $C_7H_{12}O_3$ | $4.7\times10^4$ | | Wang et al. (2017) | Q | 80, 239 |
| PHVXTLWKDLVJPT-UHFFFAOYSA-N | $2.7\times10^5$ | | Wang et al. (2017) | Q | 80, 240 |
| MCM:C624CO2H | $3.6\times10^4$ | | Wang et al. (2017) | Q | 80, 238 |
| $C_7H_{12}O_3$ | $3.0\times10^4$ | | Wang et al. (2017) | Q | 80, 239 |
| IOGKLHPXZHQWIC-UHFFFAOYSA-N | $4.3\times10^5$ | | Wang et al. (2017) | Q | 80, 240 |
| MCM:H3M3C5CO2H | $1.1\times10^4$ | | Wang et al. (2017) | Q | 80, 238 |
| $C_7H_{14}O_3$ | $1.7\times10^4$ | | Wang et al. (2017) | Q | 80, 239 |
| BHWOTLCMMUBASI-UHFFFAOYSA-N | $2.8\times10^5$ | | Wang et al. (2017) | Q | 80, 240 |
| MCM:C811OH | $3.6\times10^4$ | | Wang et al. (2017) | Q | 80, 238 |
| $C_8H_{14}O_3$ | $1.7\times10^6$ | | Wang et al. (2017) | Q | 80, 239 |
| UVNHICQQCWWSMK-UHFFFAOYSA-N | $9.8\times10^5$ | | Wang et al. (2017) | Q | 80, 240 |
| MCM:C812OH | $3.4\times10^6$ | | Wang et al. (2017) | Q | 80, 238 |
| $C_8H_{14}O_4$ | $2.9\times10^8$ | | Wang et al. (2017) | Q | 80, 239 |
| VFUDATRQVLCLSF-UHFFFAOYSA-N | $1.2\times10^7$ | | Wang et al. (2017) | Q | 80, 240 |
| MCM:C812OOH | $1.8\times10^9$ | | Wang et al. (2017) | Q | 80, 238 |
| $C_8H_{14}O_5$ | $4.9\times10^8$ | | Wang et al. (2017) | Q | 80, 239 |
| TXANCYYWZQZPGK-UHFFFAOYSA-N | $1.5\times10^8$ | | Wang et al. (2017) | Q | 80, 240 |
| MCM:C823OH | $3.3\times10^4$ | | Wang et al. (2017) | Q | 80, 238 |
| $C_8H_{14}O_3$ | $2.0\times10^6$ | | Wang et al. (2017) | Q | 80, 239 |
| HTVBHPNLYICQSI-UHFFFAOYSA-N | $9.3\times10^4$ | | Wang et al. (2017) | Q | 80, 240 |
| MCM:C825OH | $5.5\times10^6$ | | Wang et al. (2017) | Q | 80, 238 |
| $C_8H_{14}O_4$ | $3.7\times10^7$ | | Wang et al. (2017) | Q | 80, 239 |
| GZVYHBJSBFFFII-UHFFFAOYSA-N | $5.9\times10^6$ | | Wang et al. (2017) | Q | 80, 240 |
| MCM:C825OOH | $2.8\times10^9$ | | Wang et al. (2017) | Q | 80, 238 |
| $C_8H_{14}O_5$ | $7.8\times10^8$ | | Wang et al. (2017) | Q | 80, 239 |
| XYVJCWMMBMHSFM-UHFFFAOYSA-N | $1.8\times10^7$ | | Wang et al. (2017) | Q | 80, 240 |
| MCM:C137OH | $3.1\times10^4$ | | Wang et al. (2017) | Q | 80, 238 |
| $C_{13}H_{22}O_3$ | $1.7\times10^6$ | | Wang et al. (2017) | Q | 80, 239 |
| GRQGZSRLPJOOBF-UHFFFAOYSA-N | $2.1\times10^5$ | | Wang et al. (2017) | Q | 80, 240 |
| MCM:C139OH | $6.0\times10^7$ | | Wang et al. (2017) | Q | 80, 238 |
| $C_{13}H_{22}O_4$ | $7.6\times10^9$ | | Wang et al. (2017) | Q | 80, 239 |
| RQGWJIBQKQBFEC-UHFFFAOYSA-N | $1.4\times10^8$ | | Wang et al. (2017) | Q | 80, 240 |
| MCM:C139OOH | $1.8\times10^9$ | | Wang et al. (2017) | Q | 80, 238 |
| $C_{13}H_{22}O_5$ | $1.7\times10^{10}$ | | Wang et al. (2017) | Q | 80, 239 |
| MQHLLQIJDGKDCT-UHFFFAOYSA-N | $9.8\times10^8$ | | Wang et al. (2017) | Q | 80, 240 |
| MCM:C44OOH | $7.3\times10^8$ | | Wang et al. (2017) | Q | 80, 238 |
| $C_4H_6O_5$ | $4.1\times10^7$ | | Wang et al. (2017) | Q | 80, 239 |
| RIVXGQNQRYOWAN-UHFFFAOYSA-N | $1.9\times10^5$ | | Wang et al. (2017) | Q | 80, 240 |





Table A3.7: Carboxylic acids (RCOOH) and peroxy carboxylic acids (RCOOOH) (...continued)

| Substance Formula (Trivial Name) [CAS Registry Number] InChIKey | $H_s^{cp}$ (at $T^{\ominus}$) $\left[\dfrac{\mathrm{mol}}{\mathrm{m^3\,Pa}}\right]$ | $\dfrac{\mathrm{d}\ln H_s^{cp}}{\mathrm{d}(1/T)}$ [K] | Reference | Type | Note |
|---|---|---|---|---|---|
| MCM:MALDALCO2H | $3.4\times10^4$ | | Wang et al. (2017) | Q | 80, 238 |
| $C_4H_4O_3$ | $6.0\times10^3$ | | Wang et al. (2017) | Q | 80, 239 |
| ZOIRMVZWDRLJPI-UHFFFAOYSA-N | $5.9\times10^3$ | | Wang et al. (2017) | Q | 80, 240 |
| MCM:PRPAL2CO2H | $9.8\times10^3$ | | Wang et al. (2017) | Q | 80, 238 |
| $C_4H_6O_3$ | $5.0\times10^3$ | | Wang et al. (2017) | Q | 80, 239 |
| VOKUMXABRRXHAR-UHFFFAOYSA-N | $2.0\times10^3$ | | Wang et al. (2017) | Q | 80, 240 |
| MCM:C3MCOCO2H | $2.3\times10^4$ | | Wang et al. (2017) | Q | 80, 238 |
| $C_5H_6O_3$ | $9.6\times10^3$ | | Wang et al. (2017) | Q | 80, 239 |
| VXAWORVMCLXEKH-UHFFFAOYSA-N | $7.8\times10^2$ | | Wang et al. (2017) | Q | 80, 240 |
| MCM:CO1M22CO2H | $5.4\times10^3$ | | Wang et al. (2017) | Q | 80, 238 |
| $C_5H_8O_3$ | $1.3\times10^3$ | | Wang et al. (2017) | Q | 80, 239 |
| SUMZWDXUXLTFFX-UHFFFAOYSA-N | $6.5\times10^2$ | | Wang et al. (2017) | Q | 80, 240 |
| MCM:MC3ODBCO2H | $2.3\times10^4$ | | Wang et al. (2017) | Q | 80, 238 |
| $C_5H_6O_3$ | $9.3\times10^3$ | | Wang et al. (2017) | Q | 80, 239 |
| QPFVAKLVWFPTSX-UHFFFAOYSA-N | $3.4\times10^3$ | | Wang et al. (2017) | Q | 80, 240 |
| MCM:C3EODBCO2H | $2.0\times10^4$ | | Wang et al. (2017) | Q | 80, 238 |
| $C_6H_8O_3$ | $8.3\times10^3$ | | Wang et al. (2017) | Q | 80, 239 |
| XIZBOUDCWHEQCF-UHFFFAOYSA-N | $4.6\times10^2$ | | Wang et al. (2017) | Q | 80, 240 |
| MCM:C3M2COCO2H | $1.6\times10^4$ | | Wang et al. (2017) | Q | 80, 238 |
| $C_6H_8O_3$ | $9.1\times10^3$ | | Wang et al. (2017) | Q | 80, 239 |
| HLFIROOBCUFLFC-UHFFFAOYSA-N | $5.8\times10^3$ | | Wang et al. (2017) | Q | 80, 240 |
| MCM:C522CO2H | $1.3\times10^4$ | | Wang et al. (2017) | Q | 80, 238 |
| $C_6H_8O_3$ | $2.3\times10^4$ | | Wang et al. (2017) | Q | 80, 239 |
| NSMTVDIDRFOALU-UHFFFAOYSA-N | $9.3\times10^3$ | | Wang et al. (2017) | Q | 80, 240 |
| MCM:RGDCO2H | $2.0\times10^4$ | | Wang et al. (2017) | Q | 80, 238 |
| $C_6H_8O_3$ | $8.5\times10^3$ | | Wang et al. (2017) | Q | 80, 239 |
| HNXWTHGWCGDLKB-UHFFFAOYSA-N | $3.0\times10^3$ | | Wang et al. (2017) | Q | 80, 240 |
| MCM:C615CO2H | $3.6\times10^6$ | | Wang et al. (2017) | Q | 80, 238 |
| $C_7H_{10}O_4$ | $5.4\times10^5$ | | Wang et al. (2017) | Q | 80, 239 |
| BRDQYLNZHVUNQF-UHFFFAOYSA-N | $2.6\times10^4$ | | Wang et al. (2017) | Q | 80, 240 |
| MCM:C722OOH | $3.0\times10^8$ | | Wang et al. (2017) | Q | 80, 238 |
| $C_7H_{12}O_5$ | $1.6\times10^7$ | | Wang et al. (2017) | Q | 80, 239 |
| XJSSOLXEKAZFKL-UHFFFAOYSA-N | $8.0\times10^5$ | | Wang et al. (2017) | Q | 80, 240 |
| MCM:IP3ODBCO2H | $1.9\times10^4$ | | Wang et al. (2017) | Q | 80, 238 |
| $C_7H_{10}O_3$ | $4.5\times10^3$ | | Wang et al. (2017) | Q | 80, 239 |
| HLXOYOZOZOGAJW-UHFFFAOYSA-N | $1.1\times10^3$ | | Wang et al. (2017) | Q | 80, 240 |
| MCM:PC3ODBCO2H | $1.7\times10^4$ | | Wang et al. (2017) | Q | 80, 238 |
| $C_7H_{10}O_3$ | $5.4\times10^3$ | | Wang et al. (2017) | Q | 80, 239 |
| SHOOYBDWMZRRKO-UHFFFAOYSA-N | $2.2\times10^3$ | | Wang et al. (2017) | Q | 80, 240 |





Table A3.7: Carboxylic acids (RCOOH) and peroxy carboxylic acids (RCOOOH) (... continued)

| Substance / Formula / (Trivial Name) / [CAS Registry Number] / InChIKey | $H_s^{cp}$ (at $T^{\ominus}$) $\left[\dfrac{\mathrm{mol}}{\mathrm{m^3\,Pa}}\right]$ | $\dfrac{\mathrm{d}\ln H_s^{cp}}{\mathrm{d}(1/T)}$ [K] | Reference | Type | Note |
|---|---|---|---|---|---|
| MCM:C721CHO | $1.2\times10^4$ | | Wang et al. (2017) | Q | 80, 238 |
| $C_8H_{12}O_3$ | $2.8\times10^5$ | | Wang et al. (2017) | Q | 80, 239 |
| MQTHUXQLMJSVGN-UHFFFAOYSA-N | $2.9\times10^4$ | | Wang et al. (2017) | Q | 80, 240 |
| MCM:C729CO2H | $1.0\times10^4$ | | Wang et al. (2017) | Q | 80, 238 |
| $C_8H_{12}O_3$ | $1.1\times10^5$ | | Wang et al. (2017) | Q | 80, 239 |
| NHNJUKWYLMFOQP-UHFFFAOYSA-N | $1.7\times10^4$ | | Wang et al. (2017) | Q | 80, 240 |
| MCM:C823CO | $1.0\times10^4$ | | Wang et al. (2017) | Q | 80, 238 |
| $C_8H_{12}O_3$ | $2.2\times10^5$ | | Wang et al. (2017) | Q | 80, 239 |
| KUOQBJGRIDTXEZ-UHFFFAOYSA-N | $7.1\times10^3$ | | Wang et al. (2017) | Q | 80, 240 |
| MCM:C822CO2H | $4.4\times10^4$ | 15000 | Wieser et al. (2023) | Q | 437 |
| $C_9H_{14}O_3$ | $8.5\times10^3$ | | Wang et al. (2017) | Q | 80, 238 |
| ADULCIYKWVJSFE-UHFFFAOYSA-N | $9.3\times10^4$ | | Wang et al. (2017) | Q | 80, 239 |
| | $3.0\times10^4$ | | Wang et al. (2017) | Q | 80, 240 |
| MCM:C830CO2H | $9.1\times10^3$ | | Wang et al. (2017) | Q | 80, 238 |
| $C_9H_{14}O_3$ | $1.9\times10^5$ | | Wang et al. (2017) | Q | 80, 239 |
| NWNQYPOMHUZLML-UHFFFAOYSA-N | $2.3\times10^4$ | | Wang et al. (2017) | Q | 80, 240 |
| MCM:C89CO2H | $9.1\times10^3$ | | Wang et al. (2017) | Q | 80, 238 |
| $C_9H_{14}O_3$ | $2.0\times10^5$ | | Wang et al. (2017) | Q | 80, 239 |
| RFGXRYLTPBBFIG-UHFFFAOYSA-N | $7.4\times10^4$ | | Wang et al. (2017) | Q | 80, 240 |
| MCM:C126CO2H | $1.0\times10^4$ | | Wang et al. (2017) | Q | 80, 238 |
| $C_{13}H_{20}O_3$ | $2.4\times10^5$ | | Wang et al. (2017) | Q | 80, 239 |
| FRZAETYQWIEERW-UHFFFAOYSA-N | $2.6\times10^5$ | | Wang et al. (2017) | Q | 80, 240 |
| MCM:C137CO | $1.0\times10^4$ | | Wang et al. (2017) | Q | 80, 238 |
| $C_{13}H_{20}O_3$ | $2.2\times10^5$ | | Wang et al. (2017) | Q | 80, 239 |
| FQUQDHNCRLRZNB-UHFFFAOYSA-N | $7.8\times10^4$ | | Wang et al. (2017) | Q | 80, 240 |
| MCM:C136CO2H | $8.9\times10^3$ | | Wang et al. (2017) | Q | 80, 238 |
| $C_{14}H_{22}O_3$ | $1.5\times10^5$ | | Wang et al. (2017) | Q | 80, 239 |
| MUFZWTUKICSBHZ-UHFFFAOYSA-N | $7.8\times10^3$ | | Wang et al. (2017) | Q | 80, 240 |
| MCM:COHM2CO2H | $2.0\times10^5$ | | Wang et al. (2017) | Q | 80, 238 |
| $C_4H_6O_4$ | $7.1\times10^5$ | | Wang et al. (2017) | Q | 80, 239 |
| DDNPKLYNPFAIEE-UHFFFAOYSA-N | $2.0\times10^2$ | | Wang et al. (2017) | Q | 80, 240 |
| MCM:C1H4C5CO2H | $1.8\times10^7$ | | Wang et al. (2017) | Q | 80, 238 |
| $C_6H_{10}O_4$ | $3.4\times10^7$ | | Wang et al. (2017) | Q | 80, 239 |
| NIVGPRREFKECGM-UHFFFAOYSA-N | $3.2\times10^6$ | | Wang et al. (2017) | Q | 80, 240 |
| MCM:ECOCO2H | $7.3\times10^3$ | | Wang et al. (2017) | Q | 80, 238 |
| $C_4H_6O_3$ | $4.1\times10^1$ | | Wang et al. (2017) | Q | 80, 239 |
| TYEYBOSBBBHJIV-UHFFFAOYSA-N | $5.8$ | | Wang et al. (2017) | Q | 80, 240 |
| MCM:C41CO2H | $6.5\times10^3$ | | Wang et al. (2017) | Q | 80, 238 |
| $C_5H_8O_3$ | $4.6\times10^3$ | | Wang et al. (2017) | Q | 80, 239 |
| GCXJINGJZAOJHR-UHFFFAOYSA-N | $3.0\times10^3$ | | Wang et al. (2017) | Q | 80, 240 |



Table A3.7: Carboxylic acids (RCOOH) and peroxy carboxylic acids (RCOOOH) (...continued)

| Substance<br>Formula<br>(Trivial Name)<br>[CAS Registry Number]<br>InChIKey | $H_s^{cp}$<br>(at $T^\ominus$)<br>$\left[\dfrac{\mathrm{mol}}{\mathrm{m}^3\,\mathrm{Pa}}\right]$ | $\dfrac{\mathrm{d}\ln H_s^{cp}}{\mathrm{d}(1/T)}$<br><br>[K] | Reference | Type | Note |
|---|---|---|---|---|---|
| MCM:C5CO14OH<br>$C_5H_6O_3$<br>XGTKSWVCNVUVHG-UHFFFAOYSA-N | $2.3\times10^4$<br>$1.6\times10^4$<br>$1.4\times10^4$ | | Wang et al. (2017)<br>Wang et al. (2017)<br>Wang et al. (2017) | Q<br>Q<br>Q | 80, 238<br>80, 239<br>80, 240 |
| MCM:CO2C4CO2H<br>$C_5H_8O_3$<br>JOOXCMJARBKPKM-UHFFFAOYSA-N | $5.6\times10^3$<br>$4.6\times10^5$<br>$7.6\times10^4$ | | Wang et al. (2017)<br>Wang et al. (2017)<br>Wang et al. (2017) | Q<br>Q<br>Q | 80, 238<br>80, 239<br>80, 240 |
| MCM:CO3C4CO2H<br>$C_5H_8O_3$<br>FHSUFDYFOHSYHI-UHFFFAOYSA-N | $5.6\times10^3$<br>$1.5\times10^4$<br>$5.4\times10^3$ | | Wang et al. (2017)<br>Wang et al. (2017)<br>Wang et al. (2017) | Q<br>Q<br>Q | 80, 238<br>80, 239<br>80, 240 |
| MCM:IPGLYOXOH<br>$C_5H_8O_3$<br>QHKABHOOEWYVLI-UHFFFAOYSA-N | $6.5\times10^3$<br>$1.7\times10^1$<br>$5.4$ | | Wang et al. (2017)<br>Wang et al. (2017)<br>Wang et al. (2017) | Q<br>Q<br>Q | 80, 238<br>80, 239<br>80, 240 |
| MCM:PGLYOXOH<br>$C_5H_8O_3$<br>KDVFRMMRZOCFLS-UHFFFAOYSA-N | $5.6\times10^3$<br>$2.0\times10^1$<br>$5.9$ | | Wang et al. (2017)<br>Wang et al. (2017)<br>Wang et al. (2017) | Q<br>Q<br>Q | 80, 238<br>80, 239<br>80, 240 |
| MCM:C51CO2H<br>$C_6H_{10}O_3$<br>NFIWUVRBASXMGK-UHFFFAOYSA-N | $5.3\times10^3$<br>$1.6\times10^5$<br>$1.9\times10^4$ | | Wang et al. (2017)<br>Wang et al. (2017)<br>Wang et al. (2017) | Q<br>Q<br>Q | 80, 238<br>80, 239<br>80, 240 |
| MCM:C5CODBCO2H<br>$C_6H_8O_3$<br>YUICRCSHEOZHST-UHFFFAOYSA-N | $1.6\times10^4$<br>$1.6\times10^4$<br>$2.1\times10^3$ | | Wang et al. (2017)<br>Wang et al. (2017)<br>Wang et al. (2017) | Q<br>Q<br>Q | 80, 238<br>80, 239<br>80, 240 |
| MCM:C5DBCOCO2H<br>$C_6H_8O_3$<br>LRRPKULXSOVRRZ-UHFFFAOYSA-N | $1.6\times10^4$<br>$1.4\times10^4$<br>$2.4\times10^3$ | | Wang et al. (2017)<br>Wang et al. (2017)<br>Wang et al. (2017) | Q<br>Q<br>Q | 80, 238<br>80, 239<br>80, 240 |
| MCM:C6DCRBBOH<br>$C_6H_8O_3$<br>VUZJJUOKMOGBDH-UHFFFAOYSA-N | $2.0\times10^4$<br>$1.4\times10^4$<br>$4.6\times10^3$ | | Wang et al. (2017)<br>Wang et al. (2017)<br>Wang et al. (2017) | Q<br>Q<br>Q | 80, 238<br>80, 239<br>80, 240 |
| MCM:CO2C54CO2H<br>$C_6H_{10}O_3$<br>UZTJTTKEYGHTNM-UHFFFAOYSA-N | $5.3\times10^3$<br>$8.7\times10^4$<br>$3.2\times10^4$ | | Wang et al. (2017)<br>Wang et al. (2017)<br>Wang et al. (2017) | Q<br>Q<br>Q | 80, 238<br>80, 239<br>80, 240 |
| MCM:CO2M33CO2H<br>$C_6H_{10}O_3$<br>CBZZWRQRPKSEQI-UHFFFAOYSA-N | $3.6\times10^3$<br>$8.5\times10^2$<br>$1.2\times10^3$ | | Wang et al. (2017)<br>Wang et al. (2017)<br>Wang et al. (2017) | Q<br>Q<br>Q | 80, 238<br>80, 239<br>80, 240 |
| MCM:CO3C5CO2H<br>$C_6H_{10}O_3$<br>CLJBDOUIEHLLEN-UHFFFAOYSA-N | $4.5\times10^3$<br>$2.6\times10^5$<br>$3.9\times10^4$ | | Wang et al. (2017)<br>Wang et al. (2017)<br>Wang et al. (2017) | Q<br>Q<br>Q | 80, 238<br>80, 239<br>80, 240 |
| MCM:C732OOH<br>$C_7H_{12}O_5$<br>ASSSOSHEUJWVOL-UHFFFAOYSA-N | $3.2\times10^8$<br>$1.8\times10^9$<br>$2.0\times10^6$ | | Wang et al. (2017)<br>Wang et al. (2017)<br>Wang et al. (2017) | Q<br>Q<br>Q | 80, 238<br>80, 239<br>80, 240 |



Table A3.7: Carboxylic acids (RCOOH) and peroxy carboxylic acids (RCOOOH) (... continued)

| Substance<br>Formula<br>(Trivial Name)<br>[CAS Registry Number]<br>InChIKey | $H_s^{cp}$<br>(at $T^\ominus$)<br>$\left[\dfrac{\mathrm{mol}}{\mathrm{m}^3\,\mathrm{Pa}}\right]$ | $\dfrac{\mathrm{d}\ln H_s^{cp}}{\mathrm{d}(1/T)}$<br><br>[K] | Reference | Type | Note |
|---|---|---|---|---|---|
| MCM:C7ADCCO2H<br>$C_7H_{10}O_3$<br>GJIWJSZYGQVZNC-UHFFFAOYSA-N | $1.0\times10^4$<br>$1.3\times10^4$<br>$6.5\times10^3$ | | Wang et al. (2017)<br>Wang et al. (2017)<br>Wang et al. (2017) | Q<br>Q<br>Q | 80, 238<br>80, 239<br>80, 240 |
| MCM:C7CDCCO2H<br>$C_7H_{10}O_3$<br>OAGXHOMWGVMDLP-UHFFFAOYSA-N | $1.4\times10^4$<br>$7.8\times10^3$<br>$1.1\times10^4$ | | Wang et al. (2017)<br>Wang et al. (2017)<br>Wang et al. (2017) | Q<br>Q<br>Q | 80, 238<br>80, 239<br>80, 240 |
| MCM:C7DCCO2H<br>$C_7H_{10}O_3$<br>LUXOYEZTEWTGMD-UHFFFAOYSA-N | $1.7\times10^4$<br>$8.5\times10^3$<br>$4.0\times10^3$ | | Wang et al. (2017)<br>Wang et al. (2017)<br>Wang et al. (2017) | Q<br>Q<br>Q | 80, 238<br>80, 239<br>80, 240 |
| MCM:C7DDCCO2H<br>$C_7H_{10}O_3$<br>MDDQAVHYGMRQBB-UHFFFAOYSA-N | $1.4\times10^4$<br>$8.5\times10^3$<br>$1.5\times10^3$ | | Wang et al. (2017)<br>Wang et al. (2017)<br>Wang et al. (2017) | Q<br>Q<br>Q | 80, 238<br>80, 239<br>80, 240 |
| MCM:CO25C6CO2H<br>$C_7H_{10}O_4$<br>WUQQVFQGWWMQLD-UHFFFAOYSA-N | $2.7\times10^6$<br>$2.2\times10^7$<br>$1.7\times10^6$ | | Wang et al. (2017)<br>Wang et al. (2017)<br>Wang et al. (2017) | Q<br>Q<br>Q | 80, 238<br>80, 239<br>80, 240 |
| MCM:IC7DCCO2H<br>$C_7H_{10}O_3$<br>TZKNMZPMBZQSMV-UHFFFAOYSA-N | $1.9\times10^4$<br>$6.9\times10^3$<br>$3.2\times10^3$ | | Wang et al. (2017)<br>Wang et al. (2017)<br>Wang et al. (2017) | Q<br>Q<br>Q | 80, 238<br>80, 239<br>80, 240 |
| MCM:C732CO3H<br>$C_8H_{12}O_6$<br>IARLEAITRRLVEZ-UHFFFAOYSA-N | $3.6\times10^9$<br>$2.2\times10^9$<br>$2.7\times10^6$ | | Wang et al. (2017)<br>Wang et al. (2017)<br>Wang et al. (2017) | Q<br>Q<br>Q | 80, 238<br>80, 239<br>80, 240 |
| MCM:KLIMONIC<br>$C_8H_{12}O_5$<br>POLIOAJLCGOOTJ-UHFFFAOYSA-N | $4.5\times10^8$<br>$3.4\times10^{10}$<br>$1.1\times10^9$ | | Wang et al. (2017)<br>Wang et al. (2017)<br>Wang et al. (2017) | Q<br>Q<br>Q | 80, 238<br>80, 239<br>80, 240 |
| MCM:C827CO2H<br>$C_9H_{14}O_4$<br>UOMMGFNPUVSKTE-UHFFFAOYSA-N | $1.4\times10^6$<br>$1.3\times10^6$<br>$4.9\times10^5$ | | Wang et al. (2017)<br>Wang et al. (2017)<br>Wang et al. (2017) | Q<br>Q<br>Q | 80, 238<br>80, 239<br>80, 240 |
| MCM:C88CO2H<br>$C_9H_{12}O_4$<br>LOFCXTWTTIFNLN-UHFFFAOYSA-N | $4.5\times10^6$<br>$6.6\times10^7$<br>$3.2\times10^6$ | | Wang et al. (2017)<br>Wang et al. (2017)<br>Wang et al. (2017) | Q<br>Q<br>Q | 80, 238<br>80, 239<br>80, 240 |
| MCM:KLIMONONIC<br>$C_9H_{14}O_4$<br>RRLLKTDJWPIWHS-UHFFFAOYSA-N | $2.0\times10^6$<br>$5.4\times10^7$<br>$8.5\times10^6$ | | Wang et al. (2017)<br>Wang et al. (2017)<br>Wang et al. (2017) | Q<br>Q<br>Q | 80, 238<br>80, 239<br>80, 240 |
| MCM:C928CO2H<br>$C_{10}H_{16}O_4$<br>SANKOOWGNYBKOU-UHFFFAOYSA-N | $1.1\times10^6$<br>$5.6\times10^7$<br>$1.4\times10^6$ | | Wang et al. (2017)<br>Wang et al. (2017)<br>Wang et al. (2017) | Q<br>Q<br>Q | 80, 238<br>80, 239<br>80, 240 |
| MCM:LIMONONIC<br>$C_{10}H_{16}O_3$<br>NJOIWWRMLFSDTM-UHFFFAOYSA-N | $5.7\times10^4$<br>$5.8\times10^3$<br>$5.9\times10^4$<br>$2.4\times10^5$ | 15000 | Wieser et al. (2023)<br>Wang et al. (2017)<br>Wang et al. (2017)<br>Wang et al. (2017) | Q<br>Q<br>Q<br>Q | 437<br>80, 238<br>80, 239<br>80, 240 |



Table A3.7: Carboxylic acids (RCOOH) and peroxy carboxylic acids (RCOOOH) (…continued)

| Substance Formula (Trivial Name) [CAS Registry Number] InChIKey | $H_s^{cp}$ (at $T^{\ominus}$) $\left[\dfrac{\mathrm{mol}}{\mathrm{m}^3\,\mathrm{Pa}}\right]$ | $\dfrac{\mathrm{d}\ln H_s^{cp}}{\mathrm{d}(1/T)}$ [K] | Reference | Type | Note |
|---|---|---|---|---|---|
| MCM:C1011CO2H $C_{11}H_{18}O_3$ IUOPXZCJHJMHNP-UHFFFAOYSA-N | $4.9\times10^3$ $1.5\times10^5$ $2.4\times10^5$ | | Wang et al. (2017) Wang et al. (2017) Wang et al. (2017) | Q Q Q | 80, 238 80, 239 80, 240 |
| MCM:C1211OOH $C_{12}H_{20}O_5$ PTZJPBFCCQIPAZ-UHFFFAOYSA-N | $3.5\times10^8$ $1.7\times10^9$ $8.1\times10^6$ | | Wang et al. (2017) Wang et al. (2017) Wang et al. (2017) | Q Q Q | 80, 238 80, 239 80, 240 |
| MCM:C1211CO2H $C_{13}H_{20}O_5$ QUEDJGCZCJPFDA-UHFFFAOYSA-N | $4.7\times10^8$ $9.3\times10^{10}$ $2.8\times10^8$ | | Wang et al. (2017) Wang et al. (2017) Wang et al. (2017) | Q Q Q | 80, 238 80, 239 80, 240 |
| MCM:C1211CO3H $C_{13}H_{20}O_6$ DVDQWECWXWLOQZ-UHFFFAOYSA-N | $3.6\times10^9$ $4.5\times10^9$ $8.1\times10^7$ | | Wang et al. (2017) Wang et al. (2017) Wang et al. (2017) | Q Q Q | 80, 238 80, 239 80, 240 |
| MCM:C131CO2H $C_{14}H_{22}O_4$ QDUXHZZYSCMHPK-UHFFFAOYSA-N | $2.2\times10^6$ $1.4\times10^8$ $5.9\times10^7$ | | Wang et al. (2017) Wang et al. (2017) Wang et al. (2017) | Q Q Q | 80, 238 80, 239 80, 240 |
| MCM:C147CO $C_{14}H_{20}O_5$ MMRJODLWTRWNRH-UHFFFAOYSA-N | $1.4\times10^9$ $3.2\times10^8$ $2.5\times10^5$ | | Wang et al. (2017) Wang et al. (2017) Wang et al. (2017) | Q Q Q | 80, 238 80, 239 80, 240 |
| MCM:C147OOH $C_{14}H_{22}O_6$ MOUQEZQXHGSYER-UHFFFAOYSA-N | $1.7\times10^{11}$ $1.0\times10^9$ $3.0\times10^8$ | | Wang et al. (2017) Wang et al. (2017) Wang et al. (2017) | Q Q Q | 80, 238 80, 239 80, 240 |
| MCM:C141CO2H $C_{15}H_{24}O_3$ VGNFSHPHIKPNED-UHFFFAOYSA-N | $5.4\times10^3$ $2.1\times10^5$ $3.9\times10^4$ | | Wang et al. (2017) Wang et al. (2017) Wang et al. (2017) | Q Q Q | 80, 238 80, 239 80, 240 |
| MCM:HOCH2COCO2H $C_3H_4O_4$ HHDDCCUIIUWNGJ-UHFFFAOYSA-N | $1.1\times10^6$ $5.3\times10^2$ $4.6\times10^3$ | | Wang et al. (2017) Wang et al. (2017) Wang et al. (2017) | Q Q Q | 80, 238 80, 239 80, 240 |
| MCM:H3C2C4CO2H $C_5H_8O_4$ IQZHXDCQZNRIRW-UHFFFAOYSA-N | $8.5\times10^5$ $2.0\times10^6$ $2.8\times10^5$ | | Wang et al. (2017) Wang et al. (2017) Wang et al. (2017) | Q Q Q | 80, 238 80, 239 80, 240 |
| MCM:HMVKBCO2H $C_5H_8O_4$ ISMQTZHDOHOVBT-UHFFFAOYSA-N | $1.7\times10^7$ $2.1\times10^6$ $6.0\times10^5$ | | Wang et al. (2017) Wang et al. (2017) Wang et al. (2017) | Q Q Q | 80, 238 80, 239 80, 240 |
| MCM:C517CO2H $C_6H_{10}O_4$ APPVMSIJNZFGHP-UHFFFAOYSA-N | $1.4\times10^7$ $1.3\times10^7$ $2.0\times10^6$ | | Wang et al. (2017) Wang et al. (2017) Wang et al. (2017) | Q Q Q | 80, 238 80, 239 80, 240 |
| MCM:C519CO2H $C_6H_{10}O_4$ ULHNCRHSCMPZGW-UHFFFAOYSA-N | $1.4\times10^7$ $3.8\times10^6$ $3.8\times10^6$ | | Wang et al. (2017) Wang et al. (2017) Wang et al. (2017) | Q Q Q | 80, 238 80, 239 80, 240 |



Table A3.7: Carboxylic acids (RCOOH) and peroxy carboxylic acids (RCOOOH) (...continued)

| Substance Formula (Trivial Name) [CAS Registry Number] InChIKey | $H_s^{cp}$ (at $T^{\ominus}$) $\left[\dfrac{\text{mol}}{\text{m}^3\,\text{Pa}}\right]$ | $\dfrac{\text{d}\ln H_s^{cp}}{\text{d}(1/T)}$ [K] | Reference | Type | Note |
|---|---|---|---|---|---|
| MCM:C732OH | $1.1\times10^7$ | | Wang et al. (2017) | Q | 80, 238 |
| $C_7H_{12}O_4$ | $4.7\times10^8$ | | Wang et al. (2017) | Q | 80, 239 |
| PKQMWBHBXJKPID-UHFFFAOYSA-N | $4.2\times10^6$ | | Wang et al. (2017) | Q | 80, 240 |
| MCM:C734CO | $8.0\times10^9$ | | Wang et al. (2017) | Q | 80, 238 |
| $C_7H_{10}O_5$ | $6.6\times10^7$ | | Wang et al. (2017) | Q | 80, 239 |
| CPJVKYXUMUTCGW-UHFFFAOYSA-N | $3.2\times10^7$ | | Wang et al. (2017) | Q | 80, 240 |
| MCM:C734OH | $1.9\times10^9$ | | Wang et al. (2017) | Q | 80, 238 |
| $C_7H_{12}O_5$ | $8.7\times10^{10}$ | | Wang et al. (2017) | Q | 80, 239 |
| RDRUHSRUZHRAFB-UHFFFAOYSA-N | $4.4\times10^6$ | | Wang et al. (2017) | Q | 80, 240 |
| MCM:C734OOH | $1.1\times10^{12}$ | | Wang et al. (2017) | Q | 80, 238 |
| $C_7H_{12}O_6$ | $2.0\times10^{11}$ | | Wang et al. (2017) | Q | 80, 239 |
| ABEMFUWOTMEDPB-UHFFFAOYSA-N | $1.3\times10^8$ | | Wang et al. (2017) | Q | 80, 240 |
| MCM:H3C25CCO2H | $4.1\times10^8$ | | Wang et al. (2017) | Q | 80, 238 |
| $C_7H_{10}O_5$ | $4.1\times10^9$ | | Wang et al. (2017) | Q | 80, 239 |
| FMZPEVNVWACALO-UHFFFAOYSA-N | $2.7\times10^7$ | | Wang et al. (2017) | Q | 80, 240 |
| MCM:C813OH | $2.1\times10^{10}$ | | Wang et al. (2017) | Q | 80, 238 |
| $C_8H_{14}O_5$ | $6.5\times10^8$ | | Wang et al. (2017) | Q | 80, 239 |
| UHMOZKUUXAVADP-UHFFFAOYSA-N | $2.0\times10^7$ | | Wang et al. (2017) | Q | 80, 240 |
| MCM:C813OOH | $6.2\times10^{11}$ | | Wang et al. (2017) | Q | 80, 238 |
| $C_8H_{14}O_6$ | $1.7\times10^8$ | | Wang et al. (2017) | Q | 80, 239 |
| FNGYAHKKJFMZCA-UHFFFAOYSA-N | $4.3\times10^7$ | | Wang et al. (2017) | Q | 80, 240 |
| MCM:C825CO | $2.2\times10^7$ | | Wang et al. (2017) | Q | 80, 238 |
| $C_8H_{12}O_4$ | $2.6\times10^5$ | | Wang et al. (2017) | Q | 80, 239 |
| PGSQVIJDIYGHPF-UHFFFAOYSA-N | $4.9\times10^4$ | | Wang et al. (2017) | Q | 80, 240 |
| MCM:HOPINONIC | $8.9\times10^5$ | | Wang et al. (2017) | Q | 80, 238 |
| $C_{10}H_{16}O_4$ | $6.0\times10^7$ | | Wang et al. (2017) | Q | 80, 239 |
| MZHKOIVHCBFXJV-UHFFFAOYSA-N | $1.0\times10^7$ | | Wang et al. (2017) | Q | 80, 240 |
| MCM:C1211OH | $1.1\times10^7$ | | Wang et al. (2017) | Q | 80, 238 |
| $C_{12}H_{20}O_4$ | $2.7\times10^8$ | | Wang et al. (2017) | Q | 80, 239 |
| JNTJQAWMLAKHDW-UHFFFAOYSA-N | $2.8\times10^7$ | | Wang et al. (2017) | Q | 80, 240 |
| MCM:C1212OH | $1.2\times10^9$ | | Wang et al. (2017) | Q | 80, 238 |
| $C_{12}H_{20}O_5$ | $4.7\times10^{11}$ | | Wang et al. (2017) | Q | 80, 239 |
| OINCQTVJDDDYTJ-UHFFFAOYSA-N | $8.9\times10^7$ | | Wang et al. (2017) | Q | 80, 240 |
| MCM:C1212OOH | $6.5\times10^{11}$ | | Wang et al. (2017) | Q | 80, 238 |
| $C_{12}H_{20}O_6$ | $1.8\times10^{11}$ | | Wang et al. (2017) | Q | 80, 239 |
| KZNQDAPBLSOIIR-UHFFFAOYSA-N | $1.8\times10^{10}$ | | Wang et al. (2017) | Q | 80, 240 |
| MCM:C1213CO | $1.5\times10^{12}$ | | Wang et al. (2017) | Q | 80, 238 |
| $C_{12}H_{18}O_6$ | $3.3\times10^{12}$ | | Wang et al. (2017) | Q | 80, 239 |
| IKFUCIVDZJUENZ-UHFFFAOYSA-N | $2.1\times10^9$ | | Wang et al. (2017) | Q | 80, 240 |





Table A3.7: Carboxylic acids (RCOOH) and peroxy carboxylic acids (RCOOOH) (...continued)

| Substance Formula (Trivial Name) [CAS Registry Number] InChIKey | $H_s^{cp}$ (at $T^{\ominus}$) $\left[\dfrac{\mathrm{mol}}{\mathrm{m}^3\,\mathrm{Pa}}\right]$ | $\dfrac{\mathrm{d}\ln H_s^{cp}}{\mathrm{d}(1/T)}$ [K] | Reference | Type | Note |
|---|---|---|---|---|---|
| MCM:C1213OH $C_{12}H_{20}O_6$ DPGLQQDGNMDAAZ-UHFFFAOYSA-N | $7.1\times10^{12}$ $9.8\times10^{11}$ $1.3\times10^{8}$ | | Wang et al. (2017) Wang et al. (2017) Wang et al. (2017) | Q Q Q | 80, 238 80, 239 80, 240 |
| MCM:C1213OOH $C_{12}H_{20}O_7$ ZUGVXCQEOPRNFT-UHFFFAOYSA-N | $1.9\times10^{14}$ $1.3\times10^{13}$ $3.9\times10^{8}$ | | Wang et al. (2017) Wang et al. (2017) Wang et al. (2017) | Q Q Q | 80, 238 80, 239 80, 240 |
| MCM:C1310CO $C_{13}H_{20}O_5$ NHYXXXJOGAKHPV-UHFFFAOYSA-N | $4.3\times10^{9}$ $3.9\times10^{11}$ $2.6\times10^{9}$ | | Wang et al. (2017) Wang et al. (2017) Wang et al. (2017) | Q Q Q | 80, 238 80, 239 80, 240 |
| MCM:C1310OH $C_{13}H_{22}O_5$ PXEJSKZGVMTSCR-UHFFFAOYSA-N | $2.0\times10^{10}$ $1.0\times10^{11}$ $1.7\times10^{9}$ | | Wang et al. (2017) Wang et al. (2017) Wang et al. (2017) | Q Q Q | 80, 238 80, 239 80, 240 |
| MCM:C1310OOH $C_{13}H_{22}O_6$ HXUMLFGRLLCICH-UHFFFAOYSA-N | $5.4\times10^{11}$ $1.1\times10^{12}$ $5.9\times10^{7}$ | | Wang et al. (2017) Wang et al. (2017) Wang et al. (2017) | Q Q Q | 80, 238 80, 239 80, 240 |
| MCM:C147OH $C_{14}H_{22}O_5$ IPYASLZXSQHCRG-UHFFFAOYSA-N | $3.3\times10^{8}$ $5.6\times10^{8}$ $7.8\times10^{7}$ | | Wang et al. (2017) Wang et al. (2017) Wang et al. (2017) | Q Q Q | 80, 238 80, 239 80, 240 |
| MCM:C151OH $C_{15}H_{26}O_5$ RRTATIWWWPYIIC-UHFFFAOYSA-N | $6.3\times10^{9}$ $2.4\times10^{10}$ $3.7\times10^{9}$ | | Wang et al. (2017) Wang et al. (2017) Wang et al. (2017) | Q Q Q | 80, 238 80, 239 80, 240 |
| MCM:C151OOH $C_{15}H_{26}O_6$ FDHNLWUJMDRLQD-UHFFFAOYSA-N | $4.7\times10^{11}$ $1.9\times10^{11}$ $2.8\times10^{10}$ | | Wang et al. (2017) Wang et al. (2017) Wang et al. (2017) | Q Q Q | 80, 238 80, 239 80, 240 |
| MCM:CO13C3CO2H $C_{4}H_{4}O_4$ YEZSWHPLZBZVLH-UHFFFAOYSA-N | $6.2\times10^{6}$ $1.5\times10^{4}$ $1.0\times10^{3}$ | | Wang et al. (2017) Wang et al. (2017) Wang et al. (2017) | Q Q Q | 80, 238 80, 239 80, 240 |
| MCM:C512CO2H $C_{6}H_{8}O_4$ CGRSCFNOTZUXCX-UHFFFAOYSA-N | $4.0\times10^{6}$ $6.3\times10^{7}$ $1.4\times10^{6}$ | | Wang et al. (2017) Wang et al. (2017) Wang et al. (2017) | Q Q Q | 80, 238 80, 239 80, 240 |
| MCM:C6COALCO2H $C_{6}H_{8}O_4$ TWNJAPRLIZAXHA-UHFFFAOYSA-N | $4.0\times10^{6}$ $2.3\times10^{7}$ $1.4\times10^{6}$ | | Wang et al. (2017) Wang et al. (2017) Wang et al. (2017) | Q Q Q | 80, 238 80, 239 80, 240 |
| MCM:C617CO2H $C_{7}H_{10}O_4$ ZXMACHDUAJJJBQ-UHFFFAOYSA-N | $2.6\times10^{6}$ $9.6\times10^{5}$ $2.4\times10^{4}$ | | Wang et al. (2017) Wang et al. (2017) Wang et al. (2017) | Q Q Q | 80, 238 80, 239 80, 240 |
| MCM:C618CO2H $C_{7}H_{10}O_4$ MXMJWVQRGMMLRI-UHFFFAOYSA-N | $2.6\times10^{6}$ $2.3\times10^{5}$ $5.8\times10^{4}$ | | Wang et al. (2017) Wang et al. (2017) Wang et al. (2017) | Q Q Q | 80, 238 80, 239 80, 240 |



Table A3.7: Carboxylic acids (RCOOH) and peroxy carboxylic acids (RCOOOH) (...continued)

| Substance Formula (Trivial Name) [CAS Registry Number] InChIKey | $H_s^{cp}$ (at $T^{\ominus}$) $\left[\dfrac{\text{mol}}{\text{m}^3\,\text{Pa}}\right]$ | $\dfrac{\mathrm{d}\ln H_s^{cp}}{\mathrm{d}(1/T)}$ [K] | Reference | Type | Note |
|---|---|---|---|---|---|
| MCM:C626CO2H | $3.7\times10^6$ | | Wang et al. (2017) | Q | 80, 238 |
| $C_7H_{10}O_4$ | $8.3\times10^7$ | | Wang et al. (2017) | Q | 80, 239 |
| JCTYMUINHXVDEP-UHFFFAOYSA-N | $1.4\times10^6$ | | Wang et al. (2017) | Q | 80, 240 |
| MCM:C732CO | $3.7\times10^6$ | | Wang et al. (2017) | Q | 80, 238 |
| $C_7H_{10}O_4$ | $5.0\times10^7$ | | Wang et al. (2017) | Q | 80, 239 |
| ATXVRLJHWCSFDB-UHFFFAOYSA-N | $2.0\times10^5$ | | Wang et al. (2017) | Q | 80, 240 |
| MCM:C718CO2H | $2.0\times10^6$ | | Wang et al. (2017) | Q | 80, 238 |
| $C_8H_{12}O_4$ | $1.6\times10^7$ | | Wang et al. (2017) | Q | 80, 239 |
| OBYKYHDGLQUNDK-UHFFFAOYSA-N | $1.0\times10^5$ | | Wang et al. (2017) | Q | 80, 240 |
| MCM:C731CO2H | $3.5\times10^6$ | | Wang et al. (2017) | Q | 80, 238 |
| $C_8H_{12}O_4$ | $2.0\times10^8$ | | Wang et al. (2017) | Q | 80, 239 |
| URWVDUFOGQNELZ-UHFFFAOYSA-N | $3.8\times10^6$ | | Wang et al. (2017) | Q | 80, 240 |
| MCM:C87CO2H | $1.7\times10^9$ | | Wang et al. (2017) | Q | 80, 238 |
| $C_9H_{12}O_5$ | $3.6\times10^8$ | | Wang et al. (2017) | Q | 80, 239 |
| GZPOIVSRXTZTQZ-UHFFFAOYSA-N | $1.7\times10^6$ | | Wang et al. (2017) | Q | 80, 240 |
| MCM:C116CO2H | $4.1\times10^6$ | | Wang et al. (2017) | Q | 80, 238 |
| $C_{12}H_{18}O_4$ | $1.8\times10^8$ | | Wang et al. (2017) | Q | 80, 239 |
| HWWVGCXKFWUOCT-UHFFFAOYSA-N | $1.3\times10^6$ | | Wang et al. (2017) | Q | 80, 240 |
| MCM:C1211CO | $4.1\times10^6$ | | Wang et al. (2017) | Q | 80, 238 |
| $C_{12}H_{18}O_4$ | $4.2\times10^7$ | | Wang et al. (2017) | Q | 80, 239 |
| VVPUYBXPESADLR-UHFFFAOYSA-N | $8.7\times10^6$ | | Wang et al. (2017) | Q | 80, 240 |
| MCM:C1210CO2H | $3.2\times10^6$ | | Wang et al. (2017) | Q | 80, 238 |
| $C_{13}H_{20}O_4$ | $2.8\times10^8$ | | Wang et al. (2017) | Q | 80, 239 |
| MERIWEKKXYNSHZ-UHFFFAOYSA-N | $8.3\times10^5$ | | Wang et al. (2017) | Q | 80, 240 |





### A3.8   Esters (RCOOR)

Table A3.8: Esters (RCOOR)

| Substance Formula (Trivial Name) [CAS Registry Number] InChIKey | $H_s^{cp}$ (at $T^{\ominus}$) $\left[\dfrac{\mathrm{mol}}{\mathrm{m}^3\,\mathrm{Pa}}\right]$ | $\dfrac{\mathrm{d}\ln H_s^{cp}}{\mathrm{d}(1/T)}$ [K] | Reference | Type | Note |
|---|---|---|---|---|---|
| 1,3-dioxolan-2-one $C_3H_4O_3$ (ethylene carbonate) [96-49-1] KMTRUDSVKNLOMY-UHFFFAOYSA-N | $3.6{\times}10^{-2}$ | | HSDB (2015) | Q | 99 |
| carbonic acid, dimethyl ester $C_3H_6O_3$ (dimethyl carbonate) [616-38-6] IEJIGPNLZYLLBP-UHFFFAOYSA-N | $1.6{\times}10^{-1}$ | 4900 | Burkholder et al. (2019) | L | |
| | $1.6{\times}10^{-1}$ | 4900 | Burkholder et al. (2015) | L | |
| | $1.5{\times}10^{-1}$ | 5000 | Brockbank (2013) | L | 1 |
| | $1.6{\times}10^{-1}$ | 4900 | Böhme et al. (2008) | M | |
| | $1.6{\times}10^{-1}$ | 5000 | Dohnal et al. (2010) | V | 1 |
| | $2.5{\times}10^{-1}$ | | Wang et al. (2017) | Q | 80, 238 |
| | $2.1{\times}10^{-1}$ | | Wang et al. (2017) | Q | 80, 239 |
| | $2.1{\times}10^{-1}$ | | Wang et al. (2017) | Q | 80, 240 |
| | $1.6{\times}10^{-2}$ | | HSDB (2015) | Q | 99 |
| dimethyl dicarbonate $C_4H_6O_5$ [4525-33-1] GZDFHIJNHHMENY-UHFFFAOYSA-N | $2.2{\times}10^{-2}$ | | HSDB (2015) | Q | 99 |
| carbonic acid, diethyl ester $C_5H_{10}O_3$ (diethyl carbonate) [105-58-8] OIFBSDVPJOWBCH-UHFFFAOYSA-N | $1.0{\times}10^{-1}$ | 6000 | Burkholder et al. (2019) | L | |
| | $1.0{\times}10^{-1}$ | 6000 | Burkholder et al. (2015) | L | |
| | $1.0{\times}10^{-1}$ | 6400 | Brockbank (2013) | L | 1 |
| | $1.1{\times}10^{-1}$ | 6100 | Böhme et al. (2008) | M | |
| | $1.1{\times}10^{-1}$ | | Duchowicz et al. (2020) | V | 186 |
| | $1.1{\times}10^{-1}$ | | HSDB (2015) | V | |
| | $1.0{\times}10^{-1}$ | 6100 | Dohnal et al. (2010) | V | 1 |
| | $1.5{\times}10^{-2}$ | | Duchowicz et al. (2020) | Q | |
| | $1.6{\times}10^{-1}$ | | Gharagheizi et al. (2012) | Q | |
| | $6.2{\times}10^{-1}$ | | Raventos-Duran et al. (2010) | Q | 242, 243 |
| | $3.9{\times}10^{-2}$ | | Raventos-Duran et al. (2010) | Q | 244 |
| | $9.9{\times}10^{-3}$ | | Raventos-Duran et al. (2010) | Q | 245 |
| | $6.9{\times}10^{-2}$ | | Hilal et al. (2008) | Q | |
| | $2.8{\times}10^{-2}$ | | Modarresi et al. (2007) | Q | 67 |
| | $1.1{\times}10^{-1}$ | | Yaws (1999) | ? | 21, 12 |
| methyl methanoate $HCOOCH_3$ (methyl formate) [107-31-3] TZIHFWKZFHZASV-UHFFFAOYSA-N | $4.1{\times}10^{-2}$ | 4000 | Burkholder et al. (2019) | L | |
| | $4.1{\times}10^{-2}$ | 4000 | Burkholder et al. (2015) | L | |
| | $4.2{\times}10^{-2}$ | 3800 | Brockbank (2013) | L | 1 |
| | $4.1{\times}10^{-2}$ | 4000 | Sander et al. (2011) | L | |
| | $4.2{\times}10^{-2}$ | 3900 | Plyasunov et al. (2004) | L | |
| | $4.1{\times}10^{-2}$ | 4000 | Kutsuna et al. (2005) | M | |
| | $4.6{\times}10^{-2}$ | | Wittig et al. (2001) | M | |
| | $4.1{\times}10^{-2}$ | | Hoff et al. (1993) | M | |
| | $3.9{\times}10^{-2}$ | 4100 | Hartkopf and Karger (1973) | M | |
| | $4.9{\times}10^{-2}$ | | Mackay et al. (2006c) | V | |





Table A3.8: Esters (RCOOR) (...continued)

| Substance<br>Formula<br>(Trivial Name)<br>[CAS Registry Number]<br>InChIKey | $H_s^{cp}$<br>(at $T^{\ominus}$)<br>$\left[\dfrac{\text{mol}}{\text{m}^3\,\text{Pa}}\right]$ | $\dfrac{\text{d}\ln H_s^{cp}}{\text{d}(1/T)}$<br><br>[K] | Reference | Type | Note |
|---|---|---|---|---|---|
| | $4.9\times10^{-2}$ | | Mackay et al. (1995) | V | |
| | $1.2\times10^{-2}$ | | Keshavarz et al. (2022) | Q | |
| | $3.8\times10^{-1}$ | | Duchowicz et al. (2020) | Q | |
| | $5.3\times10^{-2}$ | | Wang et al. (2017) | Q | 80, 238 |
| | $1.0\times10^{-1}$ | | Wang et al. (2017) | Q | 80, 239 |
| | $8.7\times10^{-2}$ | | Wang et al. (2017) | Q | 80, 240 |
| | $4.4\times10^{-2}$ | | Li et al. (2014) | Q | 241 |
| | $5.4\times10^{-2}$ | | Gharagheizi et al. (2012) | Q | |
| | $3.9\times10^{-2}$ | | Raventos-Duran et al. (2010) | Q | 242, 243 |
| | $7.8\times10^{-2}$ | | Raventos-Duran et al. (2010) | Q | 244 |
| | $3.9\times10^{-2}$ | | Raventos-Duran et al. (2010) | Q | 245 |
| | $5.8\times10^{-2}$ | | Hilal et al. (2008) | Q | |
| | $5.8\times10^{-2}$ | | Modarresi et al. (2007) | Q | 67 |
| | | 4100 | Kühne et al. (2005) | Q | |
| | $4.6\times10^{-2}$ | | Yaffe et al. (2003) | Q | 248, 249 |
| | $3.4\times10^{-2}$ | | English and Carroll (2001) | Q | 230, 231 |
| | $2.9\times10^{-2}$ | | Katritzky et al. (1998) | Q | |
| | $8.0\times10^{-2}$ | | Suzuki et al. (1992) | Q | 232 |
| | $6.4\times10^{-2}$ | | Nirmalakhandan and Speece (1988) | Q | |
| | $4.4\times10^{-2}$ | | Duchowicz et al. (2020) | ? | 185, 21 |
| | | 4200 | Kühne et al. (2005) | ? | |
| | $5.2\times10^{-3}$ | | Yaws (1999) | ? | 21, 12 |
| | $4.4\times10^{-2}$ | | Betterton (1992) | ? | 497 |
| | $4.4\times10^{-2}$ | | Abraham et al. (1990) | ? | |
| | $4.4\times10^{-2}$ | | Hine and Mookerjee (1975) | ? | 497 |
| ethyl methanoate<br>$HCOOC_2H_5$<br>(ethyl formate)<br>[109-94-4]<br>WBJINCZRORDGAQ-UHFFFAOYSA-N | $3.4\times10^{-2}$ | 4600 | Burkholder et al. (2019) | L | |
| | $3.4\times10^{-2}$ | 4600 | Burkholder et al. (2015) | L | |
| | $3.3\times10^{-2}$ | 4000 | Brockbank (2013) | L | 1 |
| | $3.4\times10^{-2}$ | 4600 | Sander et al. (2011) | L | |
| | $3.5\times10^{-2}$ | 4600 | Plyasunov et al. (2004) | L | |
| | $3.4\times10^{-2}$ | 4600 | Kutsuna et al. (2005) | M | |
| | $4.0\times10^{-2}$ | | Wittig et al. (2001) | M | |
| | $2.3\times10^{-2}$ | | Richon et al. (1985) | M | 38 |
| | $1.9\times10^{-3}$ | 4600 | Hartkopf and Karger (1973) | M | |
| | $4.9\times10^{-2}$ | | Mackay et al. (2006c) | V | |
| | $4.9\times10^{-2}$ | | Mackay et al. (1995) | V | |
| | $3.1\times10^{-2}$ | | Abraham (1984) | V | |
| | $3.5\times10^{-2}$ | | Hine and Mookerjee (1975) | V | |
| | $1.6\times10^{-2}$ | | Keshavarz et al. (2022) | Q | |
| | $1.8\times10^{-1}$ | | Duchowicz et al. (2020) | Q | 184 |
| | $4.1\times10^{-2}$ | | Wang et al. (2017) | Q | 80, 238 |
| | $4.8\times10^{-2}$ | | Wang et al. (2017) | Q | 80, 239 |
| | $5.0\times10^{-2}$ | | Wang et al. (2017) | Q | 80, 240 |
| | $4.7\times10^{-2}$ | | Gharagheizi et al. (2012) | Q | |
| | $3.1\times10^{-2}$ | | Raventos-Duran et al. (2010) | Q | 271, 243 |
| | $3.9\times10^{-2}$ | | Raventos-Duran et al. (2010) | Q | 244 |
| | $3.1\times10^{-2}$ | | Raventos-Duran et al. (2010) | Q | 245 |



Table A3.8: Esters (RCOOR) (...continued)

| Substance Formula (Trivial Name) [CAS Registry Number] InChIKey | $H_s^{cp}$ (at $T^{\ominus}$) $\left[\dfrac{\mathrm{mol}}{\mathrm{m}^3\,\mathrm{Pa}}\right]$ | $\dfrac{\mathrm{d}\ln H_s^{cp}}{\mathrm{d}(1/T)}$ [K] | Reference | Type | Note |
|---|---|---|---|---|---|
| | $3.1\times10^{-2}$ | | Hilal et al. (2008) | Q | |
| | $5.5\times10^{-2}$ | | Modarresi et al. (2007) | Q | 67 |
| | $2.7\times10^{-2}$ | | Yaffe et al. (2003) | Q | 248, 249 |
| | $2.6\times10^{-2}$ | | English and Carroll (2001) | Q | 230, 231 |
| | $3.5\times10^{-2}$ | | Katritzky et al. (1998) | Q | |
| | $6.2\times10^{-2}$ | | Suzuki et al. (1992) | Q | 232 |
| | $5.7\times10^{-2}$ | | Nirmalakhandan and Speece (1988) | Q | |
| | $2.6\times10^{-2}$ | | Duchowicz et al. (2020) | ? | 185, 21 |
| | $4.5\times10^{-2}$ | | Yaws (1999) | ? | 21, 28 |
| | $1.7\times10^{-2}$ | | Abraham and Weathersby (1994) | ? | 21 |
| | $1.4\times10^{-3}$ | | Hoff et al. (1993) | ? | 21 |
| | $3.1\times10^{-2}$ | | Abraham et al. (1990) | ? | |
| propyl methanoate $HCOOC_3H_7$ (propyl formate) [110-74-7] KFNNIILCVOLYIR-UHFFFAOYSA-N | $2.6\times10^{-2}$ | 5100 | Burkholder et al. (2019) | L | |
| | $2.6\times10^{-2}$ | 5100 | Burkholder et al. (2015) | L | |
| | $3.1\times10^{-2}$ | 4800 | Brockbank (2013) | L | 1 |
| | $2.6\times10^{-2}$ | 5100 | Sander et al. (2011) | L | |
| | $2.8\times10^{-2}$ | 4900 | Plyasunov et al. (2004) | L | |
| | $2.6\times10^{-2}$ | 5100 | Kutsuna et al. (2005) | M | |
| | $2.3\times10^{-2}$ | | Duchowicz et al. (2020) | V | 186 |
| | $2.1\times10^{-2}$ | | Mackay et al. (2006c) | V | |
| | $2.7\times10^{-2}$ | | Hine and Mookerjee (1975) | V | |
| | $2.7\times10^{-2}$ | | Yaws (2003) | X | 237, 87 |
| | $2.1\times10^{-1}$ | | Duchowicz et al. (2020) | Q | |
| | $2.1\times10^{-2}$ | | Gharagheizi et al. (2012) | Q | |
| | $2.5\times10^{-2}$ | | Raventos-Duran et al. (2010) | Q | 242, 243 |
| | $2.0\times10^{-2}$ | | Raventos-Duran et al. (2010) | Q | 244 |
| | $2.5\times10^{-2}$ | | Raventos-Duran et al. (2010) | Q | 245 |
| | $2.9\times10^{-2}$ | | Gharagheizi et al. (2010) | Q | 246 |
| | $2.3\times10^{-2}$ | | Hilal et al. (2008) | Q | |
| | $4.5\times10^{-2}$ | | Modarresi et al. (2007) | Q | 67 |
| | $1.3\times10^{-2}$ | | Yaffe et al. (2003) | Q | 248, 272 |
| | $1.9\times10^{-2}$ | | English and Carroll (2001) | Q | 230, 231 |
| | $3.5\times10^{-2}$ | | Katritzky et al. (1998) | Q | |
| | $4.7\times10^{-2}$ | | Suzuki et al. (1992) | Q | 232 |
| | $4.4\times10^{-2}$ | | Nirmalakhandan and Speece (1988) | Q | |
| | $2.3\times10^{-2}$ | | Yaws (1999) | ? | 21, 87 |
| | $2.7\times10^{-2}$ | | Abraham et al. (1990) | ? | |
| isopropyl methanoate $HCOOC_3H_7$ (isopropyl formate) [625-55-8] RMOUBSOVHSONPZ-UHFFFAOYSA-N | $1.9\times10^{-2}$ | | Plyasunov et al. (2004) | L | |
| | $1.3\times10^{-2}$ | | Duchowicz et al. (2020) | V | 186 |
| | $1.2\times10^{-2}$ | | Hine and Mookerjee (1975) | V | |
| | $8.2\times10^{-2}$ | | Duchowicz et al. (2020) | Q | |
| | $4.4\times10^{-2}$ | | Wang et al. (2017) | Q | 80, 238 |
| | $3.1\times10^{-2}$ | | Wang et al. (2017) | Q | 80, 239 |
| | $7.8\times10^{-2}$ | | Wang et al. (2017) | Q | 80, 240 |
| | $4.1\times10^{-2}$ | | Gharagheizi et al. (2012) | Q | |
| | $2.1\times10^{-2}$ | | Hilal et al. (2008) | Q | |



Table A3.8: Esters (RCOOR) (...continued)

| Substance Formula (Trivial Name) [CAS Registry Number] InChIKey | $H_s^{cp}$ (at $T^{\ominus}$) $\left[\dfrac{\text{mol}}{\text{m}^3\,\text{Pa}}\right]$ | $\dfrac{\text{d}\ln H_s^{cp}}{\text{d}(1/T)}$ [K] | Reference | Type | Note |
|---|---|---|---|---|---|
| | $1.3\times10^{-2}$ | | Yaffe et al. (2003) | Q | 248, 249 |
| | $2.0\times10^{-2}$ | | English and Carroll (2001) | Q | 230, 274 |
| | $3.8\times10^{-2}$ | | Katritzky et al. (1998) | Q | |
| | $4.3\times10^{-2}$ | | Suzuki et al. (1992) | Q | 232 |
| | $3.9\times10^{-2}$ | | Nirmalakhandan and Speece (1988) | Q | |
| | $1.2\times10^{-2}$ | | Abraham et al. (1990) | ? | |
| butyl methanoate HCOOC$_4$H$_9$ (butyl formate) [592-84-7] NMJJFJNHVMGPGM-UHFFFAOYSA-N | $2.6\times10^{-2}$ | 3700 | Brockbank (2013) | L | 1 |
| | $1.9\times10^{-2}$ | | Plyasunov et al. (2004) | L | |
| | $1.9\times10^{-2}$ | | Wittig et al. (2001) | M | |
| | $1.9\times10^{-2}$ | | Duchowicz et al. (2020) | V | 186 |
| | $1.7\times10^{-2}$ | | Yaws (2003) | X | 237, 402 |
| | $2.3\times10^{-1}$ | | Duchowicz et al. (2020) | Q | |
| | $3.0\times10^{-2}$ | | Wang et al. (2017) | Q | 80, 238 |
| | $2.1\times10^{-2}$ | | Wang et al. (2017) | Q | 80, 239 |
| | $3.5\times10^{-2}$ | | Wang et al. (2017) | Q | 80, 240 |
| | $1.4\times10^{-2}$ | | Gharagheizi et al. (2012) | Q | |
| | $2.0\times10^{-2}$ | | Raventos-Duran et al. (2010) | Q | 242, 243 |
| | $1.6\times10^{-2}$ | | Raventos-Duran et al. (2010) | Q | 244 |
| | $1.6\times10^{-2}$ | | Raventos-Duran et al. (2010) | Q | 245 |
| | $1.9\times10^{-2}$ | | Gharagheizi et al. (2010) | Q | 246 |
| | $4.1\times10^{-2}$ | | Modarresi et al. (2007) | Q | 67 |
| | $2.0\times10^{-2}$ | | Yaffe et al. (2003) | Q | 248, 249 |
| | $3.5\times10^{-2}$ | | Katritzky et al. (1998) | Q | |
| | $1.9\times10^{-2}$ | | Yaws (1999) | ? | 21, 402 |
| (1-methylpropyl)-methanoate C$_5$H$_{10}$O$_2$ (sec-butyl formate) [589-40-2] OAEQYDZVVPONKW-UHFFFAOYSA-N | $1.9\times10^{-2}$ | | Gharagheizi et al. (2012) | Q | |
| (2-methylpropyl)-methanoate HCOOC$_4$H$_9$ (isobutyl formate) [542-55-2] AVMSWPWPYJVYKY-UHFFFAOYSA-N | $1.6\times10^{-2}$ | 5200 | Brockbank (2013) | L | 1 |
| | $1.6\times10^{-2}$ | 5200 | Plyasunov et al. (2004) | L | |
| | $1.9\times10^{-2}$ | | Duchowicz et al. (2020) | V | 186 |
| | $1.8\times10^{-2}$ | | Mackay et al. (2006c) | V | |
| | $1.8\times10^{-2}$ | | Mackay et al. (1995) | V | |
| | $1.7\times10^{-2}$ | | Hine and Mookerjee (1975) | V | |
| | $1.8\times10^{-2}$ | | Yaws (2003) | X | 237 |
| | $9.1\times10^{-2}$ | | Duchowicz et al. (2020) | Q | |
| | $9.5\times10^{-3}$ | | Gharagheizi et al. (2012) | Q | |
| | $2.1\times10^{-2}$ | | Gharagheizi et al. (2010) | Q | 246 |
| | $2.0\times10^{-2}$ | | Hilal et al. (2008) | Q | |
| | $3.3\times10^{-2}$ | | Modarresi et al. (2007) | Q | 67 |
| | $2.0\times10^{-2}$ | | Yaffe et al. (2003) | Q | 248, 249 |
| | $1.5\times10^{-2}$ | | English and Carroll (2001) | Q | 230, 260 |
| | $3.9\times10^{-2}$ | | Katritzky et al. (1998) | Q | |
| | $3.3\times10^{-2}$ | | Suzuki et al. (1992) | Q | 232 |
| | $3.1\times10^{-2}$ | | Nirmalakhandan and Speece (1988) | Q | |



Table A3.8: Esters (RCOOR) (...continued)

| Substance<br>Formula<br>(Trivial Name)<br>[CAS Registry Number]<br>InChIKey | $H_s^{cp}$<br>(at $T^{\ominus}$)<br>$\left[\dfrac{\text{mol}}{\text{m}^3\,\text{Pa}}\right]$ | $\dfrac{\text{d}\ln H_s^{cp}}{\text{d}(1/T)}$<br><br>[K] | Reference | Type | Note |
|---|---|---|---|---|---|
| | $3.3\times10^{-2}$ | | Yaws (1999) | ? | 21 |
| | $1.7\times10^{-2}$ | | Abraham et al. (1990) | ? | |
| (1,1-dimethylethyl)-methanoate | $1.4\times10^{-2}$ | 3600 | Brockbank (2013) | L | 1 |
| HCOOC$_4$H$_9$ | $1.4\times10^{-2}$ | 3600 | Arp and Schmidt (2004) | M | |
| (*tert*-butyl formate; TBF) | $2.5\times10^{-2}$ | | Yaws (2003) | X | 258 |
| [762-75-4] | $2.5\times10^{-2}$ | | Yaws (2003) | X | 237 |
| RUPAXCPQAAOIPB-UHFFFAOYSA-N | $7.3\times10^{-2}$ | | Dupeux et al. (2022) | Q | 259 |
| | $2.9\times10^{-2}$ | | Keshavarz et al. (2022) | Q | |
| | $4.0\times10^{-2}$ | | Duchowicz et al. (2020) | Q | 299 |
| | $2.3\times10^{-2}$ | | Wang et al. (2017) | Q | 80, 238 |
| | $1.1\times10^{-2}$ | | Wang et al. (2017) | Q | 80, 239 |
| | $6.6\times10^{-2}$ | | Wang et al. (2017) | Q | 80, 240 |
| | $4.5\times10^{-2}$ | | Gharagheizi et al. (2012) | Q | |
| | $2.6\times10^{-2}$ | | Gharagheizi et al. (2010) | Q | 246 |
| | $1.4\times10^{-2}$ | | Duchowicz et al. (2020) | ? | 185, 21 |
| pentyl methanoate | $1.7\times10^{-2}$ | 6000 | Brockbank (2013) | L | 1 |
| C$_6$H$_{12}$O$_2$ | $1.6\times10^{-2}$ | 5800 | Plyasunov et al. (2004) | L | |
| (pentyl formate) | $1.5\times10^{-2}$ | | Yaws (2003) | X | 237 |
| [638-49-3] | $1.1\times10^{-2}$ | | Gharagheizi et al. (2012) | Q | |
| DIQMPQMYFZXDAX-UHFFFAOYSA-N | $1.3\times10^{-2}$ | | Gharagheizi et al. (2010) | Q | 246 |
| | $1.3\times10^{-2}$ | | Hilal et al. (2008) | Q | |
| | $7.5\times10^{-3}$ | | Yaffe et al. (2003) | Q | 248, 249 |
| | $2.8\times10^{-2}$ | | Nirmalakhandan et al. (1997) | Q | |
| | $2.5\times10^{-2}$ | | Yaws (1999) | ? | 21 |
| 3-pentyl methanoate | $1.2\times10^{-2}$ | | Gharagheizi et al. (2012) | Q | |
| C$_6$H$_{12}$O$_2$ | | | | | |
| (3-pentyl formate) | | | | | |
| [58368-67-5] | | | | | |
| YMJOAYHERILFIM-UHFFFAOYSA-N | | | | | |
| 1,1-dimethylpropyl methanoate | $1.7\times10^{-2}$ | | Yaws (2003) | X | 237 |
| C$_6$H$_{12}$O$_2$ | $3.0\times10^{-2}$ | | Gharagheizi et al. (2012) | Q | |
| (1,1-dimethylpropyl formate) | $1.6\times10^{-2}$ | | Gharagheizi et al. (2010) | Q | 246 |
| [757-88-0] | | | | | |
| SVZIJXUUTNJSEJ-UHFFFAOYSA-N | | | | | |
| 1,2-dimethylpropyl methanoate | $1.7\times10^{-2}$ | | Yaws (2003) | X | 237 |
| C$_6$H$_{12}$O$_2$ | $1.1\times10^{-2}$ | | Gharagheizi et al. (2012) | Q | |
| (1,2-dimethylpropyl formate) | $1.7\times10^{-2}$ | | Gharagheizi et al. (2010) | Q | 246 |
| [66794-46-5] | | | | | |
| XEQSPMCNFJQDHU-UHFFFAOYSA-N | | | | | |
| 1-methylbutyl methanoate | $1.7\times10^{-2}$ | | Yaws (2003) | X | 237 |
| C$_6$H$_{12}$O$_2$ | $1.7\times10^{-2}$ | | Gharagheizi et al. (2012) | Q | |
| (1-methylbutyl formate) | $1.5\times10^{-2}$ | | Gharagheizi et al. (2010) | Q | 246 |
| [58368-66-4] | | | | | |
| JNRQSKFTIYCDIP-UHFFFAOYSA-N | | | | | |





Table A3.8: Esters (RCOOR) (...continued)

| Substance Formula (Trivial Name) [CAS Registry Number] InChIKey | $H_s^{cp}$ (at $T^{\ominus}$) $\left[\dfrac{\text{mol}}{\text{m}^3\,\text{Pa}}\right]$ | $\dfrac{\text{d}\ln H_s^{cp}}{\text{d}(1/T)}$ [K] | Reference | Type | Note |
|---|---|---|---|---|---|
| 2,2-dimethylpropyl methanoate $C_6H_{12}O_2$ (2,2-dimethylpropyl formate) [23361-67-3] DGMIPKNXUDSQGI-UHFFFAOYSA-N | $1.5\times10^{-2}$ $3.1\times10^{-3}$ $1.5\times10^{-2}$ | | Yaws (2003) Gharagheizi et al. (2012) Gharagheizi et al. (2010) | X Q Q | 237 246 |
| 2-methylbutyl methanoate $C_6H_{12}O_2$ (2-methylbutyl formate) [35073-27-9] DWORILFBIRYUDC-UHFFFAOYSA-N | $1.6\times10^{-2}$ $5.2\times10^{-3}$ $1.4\times10^{-2}$ | | Yaws (2003) Gharagheizi et al. (2012) Gharagheizi et al. (2010) | X Q Q | 237 246 |
| 3-methylbutyl methanoate $HCOOC_5H_{11}$ (isoamyl formate) [110-45-2] XKYICAQFSCFURC-UHFFFAOYSA-N | $1.2\times10^{-2}$ $1.5\times10^{-2}$ $1.2\times10^{-2}$ $3.9\times10^{-2}$ $9.7\times10^{-2}$ $8.5\times10^{-3}$ $1.5\times10^{-2}$ $1.7\times10^{-2}$ $3.3\times10^{-2}$ $1.6\times10^{-2}$ $1.1\times10^{-2}$ $3.4\times10^{-2}$ $2.5\times10^{-2}$ $2.4\times10^{-2}$ $1.5\times10^{-2}$ $1.5\times10^{-2}$ | 5700 | Plyasunov et al. (2004) Hine and Mookerjee (1975) Yaws (2003) Keshavarz et al. (2022) Duchowicz et al. (2020) Gharagheizi et al. (2012) Gharagheizi et al. (2010) Hilal et al. (2008) Modarresi et al. (2007) Yaffe et al. (2003) English and Carroll (2001) Katritzky et al. (1998) Suzuki et al. (1992) Nirmalakhandan and Speece (1988) Duchowicz et al. (2020) Abraham et al. (1990) | L V X Q Q Q Q Q Q Q Q Q Q Q ? ? | 237 299 246 67 248, 249 230, 231 232 185, 21 |
| hexyl methanoate $C_7H_{14}O_2$ (hexyl formate) [629-33-4] OUGPMNMLWKSBRI-UHFFFAOYSA-N | $1.0\times10^{-2}$ $1.0\times10^{-2}$ $8.0\times10^{-3}$ $8.2\times10^{-3}$ $1.1\times10^{-2}$ $1.6\times10^{-2}$ | | Plyasunov et al. (2004) Yaws (2003) Gharagheizi et al. (2012) Gharagheizi et al. (2010) Hilal et al. (2008) Nirmalakhandan et al. (1997) | L X Q Q Q Q | 237 246 |
| heptyl methanoate $C_8H_{16}O_2$ (heptyl formate) [112-23-2] XEAMDSXSXYAICO-UHFFFAOYSA-N | $6.7\times10^{-3}$ $5.9\times10^{-3}$ $5.2\times10^{-3}$ | | Yaws (2003) Gharagheizi et al. (2012) Gharagheizi et al. (2010) | X Q Q | 237 246 |
| octyl methanoate $C_9H_{18}O_2$ (octyl formate) [112-32-3] AVBRYQRTMPHARE-UHFFFAOYSA-N | $1.7\times10^{-3}$ $3.3\times10^{-3}$ $5.1\times10^{-3}$ | | Yaws (2003) Gharagheizi et al. (2010) Yaws (1999) | X Q ? | 237 246 21 |



Table A3.8: Esters (RCOOR) (. . . continued)

| Substance / Formula / (Trivial Name) / [CAS Registry Number] / InChIKey | $H_s^{cp}$ (at $T^{\ominus}$) $\left[\dfrac{\text{mol}}{\text{m}^3\,\text{Pa}}\right]$ | $\dfrac{\text{d}\ln H_s^{cp}}{\text{d}(1/T)}$ [K] | Reference | Type | Note |
|---|---|---|---|---|---|
| nonyl methanoate $C_{10}H_{20}O_2$ (nonyl formate) [5451-92-3] MOGJOLMSQWGXPA-UHFFFAOYSA-N | $2.3\times10^{-3}$ $3.9\times10^{-3}$ $2.1\times10^{-3}$ | | Yaws (2003) Gharagheizi et al. (2012) Gharagheizi et al. (2010) | X Q Q | 237 246 |
| decyl methanoate $C_{11}H_{22}O_2$ (decyl formate) [5451-52-5] BCLJZFLDSCTULJ-UHFFFAOYSA-N | $1.5\times10^{-3}$ $4.8\times10^{-3}$ $1.4\times10^{-3}$ | | Yaws (2003) Gharagheizi et al. (2012) Gharagheizi et al. (2010) | X Q Q | 237 246 |
| undecyl methanoate $C_{12}H_{24}O_2$ (undecyl formate) [5454-24-0] OASFNORBKVGDRW-UHFFFAOYSA-N | $9.3\times10^{-4}$ $6.0\times10^{-3}$ $9.3\times10^{-4}$ | | Yaws (2003) Gharagheizi et al. (2012) Gharagheizi et al. (2010) | X Q Q | 237 246 |
| dodecyl methanoate $C_{13}H_{26}O_2$ (dodecyl formate) [28303-42-6] WPSGFSPBRBRLIQ-UHFFFAOYSA-N | $6.2\times10^{-4}$ $6.6\times10^{-4}$ | | Yaws (2003) Gharagheizi et al. (2010) | X Q | 237 246 |
| vinyl methanoate $C_3H_4O_2$ (vinyl formate) [692-45-5] GFJVXXWOPWLRNU-UHFFFAOYSA-N | $5.2\times10^{-2}$ $5.2\times10^{-2}$ $1.2\times10^{-2}$ $5.3\times10^{-2}$ | | Yaws (2003) Yaws (2003) Dupeux et al. (2022) Gharagheizi et al. (2010) | X X Q Q | 258 237 259 246 |
| benzyl methanoate $C_8H_8O_2$ (benzyl formate) [104-57-4] UYWQUFXKFGHYNT-UHFFFAOYSA-N | 2.0 | 7100 | Brockbank (2013) | L | 1 |
| methyl ethanoate $CH_3COOCH_3$ (methyl acetate) [79-20-9] KXKVLQRXCPHEJC-UHFFFAOYSA-N | $8.5\times10^{-2}$ $2.0\times10^{-2}$ $8.5\times10^{-2}$ $8.1\times10^{-2}$ $8.1\times10^{-2}$ $1.2\times10^{-1}$ $6.6\times10^{-2}$ $8.3\times10^{-2}$ $4.2\times10^{-2}$ $7.7\times10^{-2}$ $7.7\times10^{-2}$ $8.6\times10^{-2}$ $1.1\times10^{-1}$ $8.7\times10^{-2}$ | 5900 5000 5900 4800 4900 7500 4500 4900 5000 4800 | Burkholder et al. (2019) Burkholder et al. (2015) Brockbank (2013) Plyasunov et al. (2004) Fenclová et al. (2014) Hiatt (2013) Arp and Schmidt (2004) Hovorka et al. (2002) Kaneko et al. (1994) Ioffe et al. (1984) Kieckbusch and King (1979b) Buttery et al. (1969) Butler and Ramchandani (1935) McKeown and Stowell (1927) | L L L L M M M M M M M M M M | 498, 1 499, 500 1 1 14 501 |





Table A3.8: Esters (RCOOR) (...continued)

| Substance Formula (Trivial Name) [CAS Registry Number] InChIKey | $H_s^{cp}$ (at $T^{\ominus}$) $\left[\dfrac{\text{mol}}{\text{m}^3\,\text{Pa}}\right]$ | $\dfrac{\text{d}\ln H_s^{cp}}{\text{d}(1/T)}$ [K] | Reference | Type | Note |
|---|---|---|---|---|---|
| | $1.1\times10^{-1}$ | | Mackay et al. (2006c) | V | |
| | $1.1\times10^{-1}$ | | Mackay et al. (1995) | V | |
| | $4.8\times10^{-2}$ | 3300 | Djerki and Laub (1988) | V | |
| | $1.1\times10^{-1}$ | 4800 | Bagno et al. (1991) | T | 473 |
| | $1.8\times10^{-1}$ | | Yaws (2003) | X | 258 |
| | $9.6\times10^{-2}$ | | Dupeux et al. (2022) | Q | 259 |
| | $1.6\times10^{-2}$ | | Keshavarz et al. (2022) | Q | |
| | $1.7\times10^{-1}$ | | Duchowicz et al. (2020) | Q | |
| | $3.4\times10^{-2}$ | | Wang et al. (2017) | Q | 80, 238 |
| | $1.2\times10^{-1}$ | | Wang et al. (2017) | Q | 80, 239 |
| | $2.5\times10^{-1}$ | | Wang et al. (2017) | Q | 80, 240 |
| | $7.3\times10^{-2}$ | | Li et al. (2014) | Q | 241 |
| | $7.8\times10^{-2}$ | | Raventos-Duran et al. (2010) | Q | 271, 243 |
| | $7.8\times10^{-2}$ | | Raventos-Duran et al. (2010) | Q | 244 |
| | $6.2\times10^{-2}$ | | Raventos-Duran et al. (2010) | Q | 245 |
| | $6.4\times10^{-2}$ | | Hilal et al. (2008) | Q | |
| | $6.0\times10^{-2}$ | | Modarresi et al. (2007) | Q | 67 |
| | | 4500 | Kühne et al. (2005) | Q | |
| | $8.6\times10^{-2}$ | | Yaffe et al. (2003) | Q | 248, 249 |
| | $5.4\times10^{-2}$ | | Yao et al. (2002) | Q | 229 |
| | $7.0\times10^{-2}$ | | English and Carroll (2001) | Q | 230, 231 |
| | $3.8\times10^{-2}$ | | Katritzky et al. (1998) | Q | |
| | $5.6\times10^{-2}$ | | Suzuki et al. (1992) | Q | 232 |
| | $3.9\times10^{-2}$ | | Nirmalakhandan and Speece (1988) | Q | |
| | $8.6\times10^{-2}$ | | Duchowicz et al. (2020) | ? | 185, 21 |
| | | 4900 | Kühne et al. (2005) | ? | |
| | $1.4\times10^{-1}$ | | Yaws (1999) | ? | 21, 12 |
| | $8.0\times10^{-2}$ | | Abraham et al. (1990) | ? | |
| ethyl ethanoate $CH_3COOC_2H_5$ (ethyl acetate) [141-78-6] XEKOWRVHYACXOJ-UHFFFAOYSA-N | $6.5\times10^{-2}$ | 5600 | Burkholder et al. (2019) | L | 502, 1 |
| | $5.9\times10^{-2}$ | 5900 | Burkholder et al. (2015) | L | |
| | $6.5\times10^{-2}$ | 5600 | Brockbank (2013) | L | 1 |
| | $5.9\times10^{-2}$ | 5900 | Sander et al. (2011) | L | |
| | $6.3\times10^{-2}$ | 5500 | Plyasunov et al. (2004) | L | |
| | $5.9\times10^{-2}$ | 5200 | Kutsuna and Kaneyasu (2021) | M | |
| | $5.2\times10^{-2}$ | 4800 | Ammari and Schroen (2019) | M | 11 |
| | $6.2\times10^{-2}$ | 5500 | Fenclová et al. (2014) | M | 1 |
| | $5.1\times10^{-2}$ | | Aprea et al. (2007) | M | |
| | $5.9\times10^{-2}$ | 5900 | Kutsuna et al. (2005) | M | |
| | $3.9\times10^{-2}$ | | van Ruth et al. (2002) | M | 14 |
| | $4.1\times10^{-2}$ | | van Ruth and Villeneuve (2002) | M | 14, 361 |
| | $3.0\times10^{-2}$ | | van Ruth et al. (2001) | M | 14 |
| | | | Dewulf et al. (1999) | M | 362 |
| | $6.6\times10^{-2}$ | | Druaux et al. (1998) | M | |
| | $3.4\times10^{-2}$ | | Welke et al. (1998) | M | |
| | $5.7\times10^{-2}$ | | Landy et al. (1995) | M | |
| | $2.8\times10^{-2}$ | | Kaneko et al. (1994) | M | 14 |
| | $4.4\times10^{-2}$ | 3900 | Kolb et al. (1992) | M | 277 |





Table A3.8: Esters (RCOOR) (...continued)

| Substance Formula (Trivial Name) [CAS Registry Number] InChIKey | $H_s^{cp}$ (at $T^{\ominus}$) $\left[\dfrac{\mathrm{mol}}{\mathrm{m^3\,Pa}}\right]$ | $\dfrac{\mathrm{d}\ln H_s^{cp}}{\mathrm{d}(1/T)}$ [K] | Reference | Type | Note |
|---|---|---|---|---|---|
| | $1.1\times10^{-1}$ | | Pividal et al. (1992) | M | 80 |
| | $4.3\times10^{-2}$ | | Guitart et al. (1989) | M | 14 |
| | $6.2\times10^{-2}$ | | Jones et al. (1988) | M | |
| | $4.3\times10^{-2}$ | | Richon et al. (1985) | M | 38 |
| | $1.3\times10^{-1}$ | | Ioffe et al. (1984) | M | 80 |
| | $5.8\times10^{-2}$ | 5300 | Kieckbusch and King (1979b) | M | 501 |
| | $5.7\times10^{-2}$ | | Nelson and Hoff (1968) | M | 297 |
| | $7.4\times10^{-2}$ | | Butler and Ramchandani (1935) | M | |
| | $7.3\times10^{-2}$ | | Mackay et al. (2006c) | V | |
| | $3.1\times10^{-2}$ | | Philippe et al. (2003) | V | 14 |
| | $7.3\times10^{-2}$ | | Mackay et al. (1995) | V | |
| | $3.6\times10^{-1}$ | | Hwang et al. (1992) | V | |
| | $7.1\times10^{-2}$ | | Yaws (2003) | X | 258 |
| | $4.7\times10^{-2}$ | 5700 | Janini and Quaddora (1986) | X | 298 |
| | $3.9\times10^{-2}$ | | Nahon et al. (2000) | C | 14 |
| | $7.8\times10^{-2}$ | | Dupeux et al. (2022) | Q | 259 |
| | $2.2\times10^{-2}$ | | Keshavarz et al. (2022) | Q | |
| | $6.9\times10^{-2}$ | | Duchowicz et al. (2020) | Q | 299 |
| | $2.8\times10^{-2}$ | | Wang et al. (2017) | Q | 80, 238 |
| | $6.2\times10^{-2}$ | | Wang et al. (2017) | Q | 80, 239 |
| | $2.5\times10^{-1}$ | | Wang et al. (2017) | Q | 80, 240 |
| | $5.2\times10^{-2}$ | | Gharagheizi et al. (2012) | Q | |
| | $6.2\times10^{-2}$ | | Raventos-Duran et al. (2010) | Q | 242, 243 |
| | $3.9\times10^{-2}$ | | Raventos-Duran et al. (2010) | Q | 244 |
| | $3.9\times10^{-2}$ | | Raventos-Duran et al. (2010) | Q | 245 |
| | $3.6\times10^{-2}$ | | Hilal et al. (2008) | Q | |
| | $4.9\times10^{-2}$ | | Modarresi et al. (2007) | Q | 67 |
| | | 4800 | Kühne et al. (2005) | Q | |
| | $7.3\times10^{-2}$ | | Yaffe et al. (2003) | Q | 248, 249 |
| | $5.4\times10^{-2}$ | | Yao et al. (2002) | Q | 229 |
| | $5.3\times10^{-2}$ | | English and Carroll (2001) | Q | 230, 231 |
| | $4.5\times10^{-2}$ | | Katritzky et al. (1998) | Q | |
| | $1.6\times10^{-1}$ | | Russell et al. (1992) | Q | 279 |
| | $4.3\times10^{-2}$ | | Suzuki et al. (1992) | Q | 232 |
| | $4.1\times10^{-2}$ | | Nirmalakhandan and Speece (1988) | Q | |
| | $7.4\times10^{-2}$ | | Duchowicz et al. (2020) | ? | 185, 21 |
| | | 5200 | Kühne et al. (2005) | ? | |
| | $7.1\times10^{-2}$ | | Yaws (1999) | ? | 21 |
| | $8.8\times10^{-2}$ | | Hoff et al. (1993) | ? | 21 |
| | $5.8\times10^{-2}$ | | Abraham et al. (1990) | ? | |
| ethyl ethanoate-1-13C $CH_3COOC_2H_5$ (ethyl acetate-1-13C) [3424-59-7] XEKOWRVHYACXOJ-AZXPZELESA-N | $7.1\times10^{-2}$ | 6500 | Hiatt (2013) | M | |



Table A3.8: Esters (RCOOR) (. . . continued)

| Substance Formula (Trivial Name) [CAS Registry Number] InChIKey | $H_s^{cp}$ (at $T^{\ominus}$) $\left[\dfrac{\mathrm{mol}}{\mathrm{m^3\,Pa}}\right]$ | $\dfrac{\mathrm{d}\ln H_s^{cp}}{\mathrm{d}(1/T)}$ [K] | Reference | Type | Note |
|---|---|---|---|---|---|
| propyl ethanoate | $4.4\times10^{-2}$ | 6100 | Burkholder et al. (2019) | L | 503, 1 |
| $CH_3COOC_3H_7$ | $4.4\times10^{-2}$ | 6000 | Brockbank (2013) | L | 1 |
| (propyl acetate) | $4.5\times10^{-2}$ | 5900 | Plyasunov et al. (2004) | L | |
| [109-60-4] | $4.5\times10^{-2}$ | 5900 | Fenclová et al. (2014) | M | 1 |
| YKYONYBAUNKHLG-UHFFFAOYSA-N | $3.9\times10^{-2}$ | | van Ruth et al. (2002) | M | 14 |
| | $3.0\times10^{-2}$ | | van Ruth and Villeneuve (2002) | M | 14, 361 |
| | $1.8\times10^{-2}$ | | van Ruth et al. (2001) | M | 14 |
| | $3.7\times10^{-2}$ | | Welke et al. (1998) | M | |
| | $2.1\times10^{-2}$ | | Kaneko et al. (1994) | M | 14 |
| | $3.1\times10^{-2}$ | | Richon et al. (1985) | M | 38 |
| | $4.5\times10^{-2}$ | 5500 | Kieckbusch and King (1979b) | M | 501 |
| | $4.6\times10^{-2}$ | | Mackay et al. (2006c) | V | |
| | $4.6\times10^{-2}$ | | Mackay et al. (1995) | V | |
| | $5.0\times10^{-2}$ | | Hine and Mookerjee (1975) | V | |
| | $5.0\times10^{-2}$ | | Butler and Ramchandani (1935) | V | |
| | $3.7\times10^{-2}$ | | Yaws (2003) | X | 258 |
| | $3.7\times10^{-2}$ | | Yaws (2003) | X | 237 |
| | $4.4\times10^{-2}$ | 6000 | Janini and Quaddora (1986) | X | 298 |
| | $5.0\times10^{-2}$ | | Dupeux et al. (2022) | Q | 259 |
| | $2.9\times10^{-2}$ | | Keshavarz et al. (2022) | Q | |
| | $7.7\times10^{-2}$ | | Duchowicz et al. (2020) | Q | |
| | $2.5\times10^{-2}$ | | Wang et al. (2017) | Q | 80, 238 |
| | $3.9\times10^{-2}$ | | Wang et al. (2017) | Q | 80, 239 |
| | $2.1\times10^{-1}$ | | Wang et al. (2017) | Q | 80, 240 |
| | $2.4\times10^{-2}$ | | Gharagheizi et al. (2012) | Q | |
| | $3.9\times10^{-2}$ | | Raventos-Duran et al. (2010) | Q | 242, 243 |
| | $3.1\times10^{-2}$ | | Raventos-Duran et al. (2010) | Q | 244 |
| | $3.1\times10^{-2}$ | | Raventos-Duran et al. (2010) | Q | 245 |
| | $4.2\times10^{-2}$ | | Gharagheizi et al. (2010) | Q | 246 |
| | $2.9\times10^{-2}$ | | Hilal et al. (2008) | Q | |
| | $4.2\times10^{-2}$ | | Modarresi et al. (2007) | Q | 67 |
| | $4.6\times10^{-2}$ | | Yaffe et al. (2003) | Q | 248, 272 |
| | $2.6\times10^{-2}$ | | Yao et al. (2002) | Q | 229 |
| | $4.0\times10^{-2}$ | | English and Carroll (2001) | Q | 230, 231 |
| | $4.1\times10^{-2}$ | | Katritzky et al. (1998) | Q | |
| | $9.9\times10^{-2}$ | | Russell et al. (1992) | Q | 279 |
| | $3.3\times10^{-2}$ | | Suzuki et al. (1992) | Q | 232 |
| | $3.3\times10^{-2}$ | | Nirmalakhandan and Speece (1988) | Q | |
| | $4.5\times10^{-2}$ | | Duchowicz et al. (2020) | ? | 185, 21 |
| | $3.4\times10^{-2}$ | | Yaws (1999) | ? | 21 |
| | $4.5\times10^{-2}$ | | Abraham et al. (1990) | ? | |
| methyl methoxyacetate $C_4H_8O_3$ [6290-49-9] QRMHDGWGLNLHMN-UHFFFAOYSA-N | 2.5 | | Hovorka et al. (2002) | M | 38 |



Table A3.8: Esters (RCOOR) (…continued)

| Substance Formula (Trivial Name) [CAS Registry Number] InChIKey | $H_s^{cp}$ (at $T^{\ominus}$) $\left[\dfrac{\text{mol}}{\text{m}^3\,\text{Pa}}\right]$ | $\dfrac{\text{d}\ln H_s^{cp}}{\text{d}(1/T)}$ [K] | Reference | Type | Note |
|---|---|---|---|---|---|
| 2-propenyl ethanoate $C_5H_8O_2$ [591-87-7] FWZUNOYOVVKUNF-UHFFFAOYSA-N | $7.6\times10^{-2}$ $2.0\times10^{-1}$ $7.8\times10^{-2}$ $3.9\times10^{-2}$ $7.0\times10^{-2}$ $8.4\times10^{-2}$ | | HSDB (2015) Raventos-Duran et al. (2010) Raventos-Duran et al. (2010) Raventos-Duran et al. (2010) Hilal et al. (2008) Modarresi et al. (2007) | V Q Q Q Q Q | 242, 243 244 245 67 |
| 2-propyl ethanoate $CH_3COOC_3H_7$ (isopropyl acetate) [108-21-4] JMMWKPVZQRWMSS-UHFFFAOYSA-N | $3.3\times10^{-2}$ $3.3\times10^{-2}$ $1.3\times10^{-2}$ $3.5\times10^{-2}$ $3.7\times10^{-2}$ $2.9\times10^{-2}$ $4.5\times10^{-2}$ $2.9\times10^{-2}$ $3.0\times10^{-2}$ $2.6\times10^{-2}$ $3.9\times10^{-2}$ $1.7\times10^{-1}$ $6.0\times10^{-2}$ $2.5\times10^{-2}$ $2.8\times10^{-2}$ $4.6\times10^{-6}$ $3.3\times10^{-2}$ $3.9\times10^{-2}$ $4.2\times10^{-2}$ $2.9\times10^{-2}$ $2.9\times10^{-2}$ $3.6\times10^{-2}$ $3.6\times10^{-2}$ $3.5\times10^{-2}$ | 6100 5600 5500 | Brockbank (2013) Plyasunov et al. (2004) Kaneko et al. (1994) Hine and Mookerjee (1975) Yaws (2003) Janini and Quaddora (1986) Dupeux et al. (2022) Keshavarz et al. (2022) Duchowicz et al. (2020) Wang et al. (2017) Wang et al. (2017) Wang et al. (2017) Gharagheizi et al. (2012) Hilal et al. (2008) Modarresi et al. (2007) Yaffe et al. (2003) Yao et al. (2002) English and Carroll (2001) Katritzky et al. (1998) Suzuki et al. (1992) Nirmalakhandan and Speece (1988) Duchowicz et al. (2020) Yaws (1999) Abraham et al. (1990) | L L M V X X Q Q Q Q Q Q Q Q Q Q Q Q Q Q Q ? ? ? | 1 14 258 298 259 184 80, 238 80, 239 80, 240 67 248, 249 229 230, 274 232 185, 21 21 |
| 2-methoxyethyl ethanoate $C_5H_{10}O_3$ (methyl cellosolve acetate) [110-49-6] XLLIQLLCWZCATF-UHFFFAOYSA-N | 3.0 9.0 | | Hovorka et al. (2002) HSDB (2015) | M V | 38 |
| glycerol monoacetate $C_5H_{10}O_4$ (acetin) [26446-35-5] KMZHZAAOEWVPSE-UHFFFAOYSA-N | $2.4\times10^4$ | | HSDB (2015) | Q | 99 |
| 1-propen-2-ol, acetate $C_5H_8O_2$ (isopropenyl acetate) [108-22-5] HETCEOQFVDFGSY-UHFFFAOYSA-N | $5.5\times10^{-3}$ | | HSDB (2015) | Q | 99 |



Table A3.8: Esters (RCOOR) (...continued)

| Substance Formula (Trivial Name) [CAS Registry Number] InChIKey | $H_s^{cp}$ (at $T^{\ominus}$) $\left[\dfrac{\text{mol}}{\text{m}^3\,\text{Pa}}\right]$ | $\dfrac{\text{d}\ln H_s^{cp}}{\text{d}(1/T)}$ [K] | Reference | Type | Note |
|---|---|---|---|---|---|
| butyl ethanoate | $3.8\times10^{-2}$ | 6300 | Brockbank (2013) | L | 1 |
| $CH_3COOC_4H_9$ | $3.5\times10^{-2}$ | 6300 | Plyasunov et al. (2004) | L | |
| (butyl acetate) | $2.4\times10^{-2}$ | | Kim and Kim (2014) | M | |
| [123-86-4] | $3.5\times10^{-2}$ | 6300 | Fenclová et al. (2014) | M | 1 |
| DKPFZGUDAPQIHT-UHFFFAOYSA-N | $2.1\times10^{-2}$ | | Helburn et al. (2008) | M | |
| | $2.8\times10^{-2}$ | | van Ruth et al. (2002) | M | 14 |
| | $2.3\times10^{-2}$ | | van Ruth and Villeneuve (2002) | M | 14, 361 |
| | $1.7\times10^{-2}$ | | van Ruth et al. (2001) | M | 14 |
| | $2.7\times10^{-2}$ | | Welke et al. (1998) | M | |
| | $1.3\times10^{-2}$ | | Kaneko et al. (1994) | M | 14 |
| | $2.3\times10^{-2}$ | 4300 | Kolb et al. (1992) | M | 277 |
| | $3.5\times10^{-2}$ | 7100 | Ioffe et al. (1984) | M | |
| | $3.5\times10^{-2}$ | 6000 | Kieckbusch and King (1979b) | M | 501 |
| | $3.2\times10^{-2}$ | | Mackay et al. (2006c) | V | |
| | $3.2\times10^{-2}$ | | Mackay et al. (1995) | V | |
| | $2.7\times10^{-2}$ | | Hwang et al. (1992) | V | |
| | $3.0\times10^{-2}$ | | Hine and Mookerjee (1975) | V | |
| | $4.2\times10^{-2}$ | | Yaws (2003) | X | 258 |
| | $3.5\times10^{-2}$ | 7500 | Janini and Quaddora (1986) | X | 298 |
| | $2.1\times10^{-2}$ | 3200 | Goldstein (1982) | X | 298 |
| | $4.3\times10^{-2}$ | | Dupeux et al. (2022) | Q | 259 |
| | $3.9\times10^{-2}$ | | Keshavarz et al. (2022) | Q | |
| | $8.2\times10^{-2}$ | | Duchowicz et al. (2020) | Q | |
| | $2.0\times10^{-2}$ | | Wang et al. (2017) | Q | 80, 238 |
| | $2.9\times10^{-2}$ | | Wang et al. (2017) | Q | 80, 239 |
| | $1.7\times10^{-1}$ | | Wang et al. (2017) | Q | 80, 240 |
| | $1.9\times10^{-2}$ | | Gharagheizi et al. (2012) | Q | |
| | $3.1\times10^{-2}$ | | Raventos-Duran et al. (2010) | Q | 242, 243 |
| | $2.0\times10^{-2}$ | | Raventos-Duran et al. (2010) | Q | 244 |
| | $2.5\times10^{-2}$ | | Raventos-Duran et al. (2010) | Q | 245 |
| | $2.3\times10^{-2}$ | | Hilal et al. (2008) | Q | |
| | $3.0\times10^{-2}$ | | Modarresi et al. (2007) | Q | 67 |
| | | 5500 | Kühne et al. (2005) | Q | |
| | $3.7\times10^{-2}$ | | Yaffe et al. (2003) | Q | 248, 249 |
| | $2.4\times10^{-2}$ | | Yao et al. (2002) | Q | 229, 267 |
| | $3.1\times10^{-2}$ | | English and Carroll (2001) | Q | 230, 231 |
| | $4.1\times10^{-2}$ | | Katritzky et al. (1998) | Q | |
| | $5.3\times10^{-2}$ | | Russell et al. (1992) | Q | 358 |
| | $1.6\times10^{-2}$ | | Suzuki et al. (1992) | Q | 232 |
| | $2.6\times10^{-2}$ | | Nirmalakhandan and Speece (1988) | Q | |
| | $3.5\times10^{-2}$ | | Duchowicz et al. (2020) | ? | 185, 21 |
| | | 5300 | Kühne et al. (2005) | ? | |
| | $3.8\times10^{-2}$ | | Yaws (1999) | ? | 21 |
| | $3.5\times10^{-2}$ | | Abraham et al. (1990) | ? | |



Table A3.8: Esters (RCOOR) (... continued)

| Substance Formula (Trivial Name) [CAS Registry Number] InChIKey | $H_s^{cp}$ (at $T^{\ominus}$) $\left[\dfrac{\text{mol}}{\text{m}^3\,\text{Pa}}\right]$ | $\dfrac{\text{d}\ln H_s^{cp}}{\text{d}(1/T)}$ [K] | Reference | Type | Note |
|---|---|---|---|---|---|
| 2-butyl ethanoate | $2.3\times10^{-2}$ | 6000 | Brockbank (2013) | L | 1 |
| $C_6H_{12}O_2$ | $2.3\times10^{-2}$ | 6200 | Plyasunov et al. (2004) | L | |
| (*sec*-butyl acetate) | $2.3\times10^{-2}$ | | HSDB (2015) | V | |
| [105-46-4] | $2.4\times10^{-2}$ | | Yaws (2003) | X | 258 |
| DCKVNWZUADLDEH-UHFFFAOYSA-N | $2.4\times10^{-2}$ | | Yaws (2003) | X | 237, 12 |
| | $2.5\times10^{-2}$ | | Dupeux et al. (2022) | Q | 259 |
| | $2.3\times10^{-2}$ | | Wang et al. (2017) | Q | 80, 238 |
| | $2.6\times10^{-2}$ | | Wang et al. (2017) | Q | 80, 239 |
| | $1.2\times10^{-1}$ | | Wang et al. (2017) | Q | 80, 240 |
| | $2.9\times10^{-2}$ | | Gharagheizi et al. (2012) | Q | |
| | $3.1\times10^{-2}$ | | Gharagheizi et al. (2010) | Q | 246 |
| | $2.9\times10^{-2}$ | | Modarresi et al. (2007) | Q | 67 |
| | $2.4\times10^{-2}$ | | Yaffe et al. (2003) | Q | 248, 249 |
| | $1.7\times10^{-2}$ | | Yao et al. (2002) | Q | 229 |
| | | | Katritzky et al. (1998) | Q | 504 |
| | $1.8\times10^{-2}$ | | Yaws (1999) | ? | 21, 12 |
| 1,1-dimethylethyl ethanoate | $1.2\times10^{-2}$ | 6100 | Brockbank (2013) | L | 1 |
| $C_6H_{12}O_2$ | $1.1\times10^{-2}$ | 5600 | Plyasunov et al. (2004) | L | |
| (*tert*-butyl acetate) | $1.1\times10^{-2}$ | | Duchowicz et al. (2020) | V | 186 |
| [540-88-5] | $3.2\times10^{-2}$ | | Yaws (2003) | X | 258 |
| WMOVHXAZOJBABW-UHFFFAOYSA-N | $3.2\times10^{-2}$ | | Yaws (2003) | X | 237 |
| | $2.6\times10^{-2}$ | | Dupeux et al. (2022) | Q | 259 |
| | $1.4\times10^{-2}$ | | Duchowicz et al. (2020) | Q | |
| | $1.4\times10^{-2}$ | | Wang et al. (2017) | Q | 80, 238 |
| | $1.6\times10^{-2}$ | | Wang et al. (2017) | Q | 80, 239 |
| | $7.4\times10^{-2}$ | | Wang et al. (2017) | Q | 80, 240 |
| | $2.4\times10^{-2}$ | | HSDB (2015) | Q | 99 |
| | $5.8\times10^{-2}$ | | Gharagheizi et al. (2012) | Q | |
| | $3.6\times10^{-2}$ | | Gharagheizi et al. (2010) | Q | 246 |
| | $2.2\times10^{-2}$ | | Modarresi et al. (2007) | Q | 67 |
| | $1.9\times10^{-2}$ | | Yao et al. (2002) | Q | 229 |
| | $2.9\times10^{-2}$ | | Yaws (1999) | ? | 21 |
| 2-methylpropyl ethanoate | $2.6\times10^{-2}$ | 6000 | Brockbank (2013) | L | 1 |
| $CH_3COOC_4H_9$ | $2.4\times10^{-2}$ | 6200 | Plyasunov et al. (2004) | L | |
| (isobutyl acetate) | $1.0\times10^{-2}$ | | Kaneko et al. (1994) | M | 14 |
| [110-19-0] | $1.9\times10^{-2}$ | | Mackay et al. (2006c) | V | |
| GJRQTCIYDGXPES-UHFFFAOYSA-N | $1.9\times10^{-2}$ | | Mackay et al. (1995) | V | |
| | $2.2\times10^{-2}$ | | Hine and Mookerjee (1975) | V | |
| | $3.7\times10^{-2}$ | | Yaws (2003) | X | 258 |
| | $2.3\times10^{-2}$ | | Yaws (2003) | X | 237 |
| | $3.7\times10^{-2}$ | | Dupeux et al. (2022) | Q | 259 |
| | $3.9\times10^{-2}$ | | Keshavarz et al. (2022) | Q | |
| | $3.2\times10^{-2}$ | | Duchowicz et al. (2020) | Q | 184 |
| | $1.0\times10^{-2}$ | | Gharagheizi et al. (2012) | Q | |
| | $4.0\times10^{-2}$ | | Gharagheizi et al. (2010) | Q | 246 |
| | $2.7\times10^{-2}$ | | Hilal et al. (2008) | Q | |





Table A3.8: Esters (RCOOR) (...continued)

| Substance / Formula / (Trivial Name) / [CAS Registry Number] / InChIKey | $H_s^{cp}$ (at $T^\ominus$) $\left[\dfrac{\text{mol}}{\text{m}^3\,\text{Pa}}\right]$ | $\dfrac{\text{d}\ln H_s^{cp}}{\text{d}(1/T)}$ [K] | Reference | Type | Note |
|---|---|---|---|---|---|
| | $3.2\times10^{-2}$ | | Modarresi et al. (2007) | Q | 67 |
| | | 5500 | Kühne et al. (2005) | Q | |
| | $1.8\times10^{-2}$ | | Yao et al. (2002) | Q | 229 |
| | $3.1\times10^{-2}$ | | English and Carroll (2001) | Q | 230, 231 |
| | $9.5\times10^{-2}$ | | Russell et al. (1992) | Q | 279 |
| | $2.3\times10^{-2}$ | | Suzuki et al. (1992) | Q | 232 |
| | $2.2\times10^{-2}$ | | Nirmalakhandan and Speece (1988) | Q | |
| | $2.2\times10^{-2}$ | | Duchowicz et al. (2020) | ? | 185, 21 |
| | | 4600 | Kühne et al. (2005) | ? | |
| | $2.3\times10^{-2}$ | | Yaws (1999) | ? | 21 |
| | $2.2\times10^{-2}$ | | Abraham et al. (1990) | ? | |
| pentyl ethanoate $CH_3COOC_5H_{11}$ (amyl acetate) [628-63-7] PGMYKACGEOXYJE-UHFFFAOYSA-N | $2.5\times10^{-2}$ | 7200 | Brockbank (2013) | L | 1 |
| | $2.7\times10^{-2}$ | 6700 | Plyasunov et al. (2004) | L | |
| | $1.3\times10^{-1}$ | 5000 | Meynier et al. (2003) | M | 38 |
| | $9.3\times10^{-3}$ | | Kaneko et al. (1994) | M | 14 |
| | $3.4\times10^{-2}$ | | Hellmann (1987) | M | 87 |
| | $2.8\times10^{-2}$ | 6500 | Kieckbusch and King (1979b) | M | 501 |
| | $2.4\times10^{-2}$ | | Mackay et al. (2006c) | V | |
| | $2.4\times10^{-2}$ | | Mackay et al. (1995) | V | |
| | $2.5\times10^{-2}$ | | Hine and Mookerjee (1975) | V | |
| | $2.8\times10^{-2}$ | | Yaws (2003) | X | 258 |
| | $2.8\times10^{-2}$ | | Yaws (2003) | X | 237 |
| | $2.5\times10^{-2}$ | | Meynier et al. (2003) | C | |
| | $3.3\times10^{-2}$ | | Dupeux et al. (2022) | Q | 259 |
| | $5.3\times10^{-2}$ | | Keshavarz et al. (2022) | Q | |
| | $8.6\times10^{-2}$ | | Duchowicz et al. (2020) | Q | |
| | $1.4\times10^{-2}$ | | Gharagheizi et al. (2012) | Q | |
| | $2.5\times10^{-2}$ | | Raventos-Duran et al. (2010) | Q | 271, 243 |
| | $2.0\times10^{-2}$ | | Raventos-Duran et al. (2010) | Q | 244 |
| | $2.0\times10^{-2}$ | | Raventos-Duran et al. (2010) | Q | 245 |
| | $2.3\times10^{-2}$ | | Gharagheizi et al. (2010) | Q | 246 |
| | $2.0\times10^{-2}$ | | Hilal et al. (2008) | Q | |
| | $2.8\times10^{-2}$ | | Modarresi et al. (2007) | Q | 67 |
| | $2.7\times10^{-2}$ | | Yaffe et al. (2003) | Q | 248, 249 |
| | $2.1\times10^{-2}$ | | Yao et al. (2002) | Q | 229, 267 |
| | $2.3\times10^{-2}$ | | English and Carroll (2001) | Q | 230, 260 |
| | $3.8\times10^{-2}$ | | Katritzky et al. (1998) | Q | |
| | $2.1\times10^{-2}$ | | Nirmalakhandan et al. (1997) | Q | |
| | $2.6\times10^{-2}$ | | Russell et al. (1992) | Q | 279 |
| | $1.9\times10^{-2}$ | | Suzuki et al. (1992) | Q | 232 |
| | $2.1\times10^{-2}$ | | Nirmalakhandan and Speece (1988) | Q | |
| | $2.3\times10^{-2}$ | | Taft et al. (1985) | Q | |
| | $2.5\times10^{-2}$ | | Duchowicz et al. (2020) | ? | 185, 21 |
| | $2.9\times10^{-2}$ | | Yaws (1999) | ? | 21 |
| | $2.8\times10^{-2}$ | | Abraham et al. (1990) | ? | |



Table A3.8: Esters (RCOOR) (...continued)

| Substance Formula (Trivial Name) [CAS Registry Number] InChIKey | $H_s^{cp}$ (at $T^{\ominus}$) $\left[\dfrac{\text{mol}}{\text{m}^3\,\text{Pa}}\right]$ | $\dfrac{\mathrm{d}\ln H_s^{cp}}{\mathrm{d}(1/T)}$ [K] | Reference | Type | Note |
|---|---|---|---|---|---|
| 2-methylbutyl ethanoate $C_7H_{14}O_2$ (2-methylbutyl acetate) [624-41-9] XHIUFYZDQBSEMF-UHFFFAOYSA-N | $2.8\times10^{-2}$ $7.4\times10^{-3}$ $2.8\times10^{-2}$ | | Yaws (2003) Gharagheizi et al. (2012) Gharagheizi et al. (2010) | X Q Q | 237 246 |
| 1,2-dimethylpropyl ethanoate $C_7H_{14}O_2$ (1,2-dimethylpropyl acetate) [5343-96-4] ZLSXRPTWWRGMTJ-UHFFFAOYSA-N | $2.9\times10^{-2}$ $1.2\times10^{-2}$ $2.9\times10^{-2}$ | | Yaws (2003) Gharagheizi et al. (2012) Gharagheizi et al. (2010) | X Q Q | 237 246 |
| 1-ethylpropyl ethanoate $C_7H_{14}O_2$ (1-ethylpropyl acetate) [620-11-1] PBKYSIMDORTIEU-UHFFFAOYSA-N | $1.3\times10^{-2}$ | | Gharagheizi et al. (2012) | Q | |
| 1,1-dimethylpropyl ethanoate $C_7H_{14}O_2$ (*tert*-pentyl acetate) [625-16-1] JCCIFDCPHCKATH-UHFFFAOYSA-N | $2.9\times10^{-2}$ $2.9\times10^{-2}$ $2.0\times10^{-2}$ $3.3\times10^{-2}$ $2.8\times10^{-2}$ | | Yaws (2003) Yaws (2003) Dupeux et al. (2022) Gharagheizi et al. (2012) Gharagheizi et al. (2010) | X X Q Q Q | 258 237 259 246 |
| 2,2-dimethylpropyl ethanoate $C_7H_{14}O_2$ (neopentyl acetate) [926-41-0] QLNYTCSELYEEPV-UHFFFAOYSA-N | $2.6\times10^{-2}$ $2.8\times10^{-2}$ | | Yaws (2003) Gharagheizi et al. (2010) | X Q | 237 246 |
| 1,2-propanediol, diacetate $C_7H_{12}O_4$ [623-84-7] MLHOXUWWKVQEJB-UHFFFAOYSA-N | $7.0\times10^{1}$ | | HSDB (2015) | Q | 99 |
| 2-pentanol, acetate $C_7H_{14}O_2$ [626-38-0] GQKZRWSUJHVIPE-UHFFFAOYSA-N | $1.2\times10^{-2}$ $1.9\times10^{-2}$ | | HSDB (2015) Gharagheizi et al. (2012) | Q Q | 99 |
| 3-methylbutyl ethanoate $CH_3COOC_5H_{11}$ (isoamyl acetate) [123-92-2] MLFHJEHSLIIPHL-UHFFFAOYSA-N | $2.1\times10^{-2}$ $2.0\times10^{-2}$ $2.2\times10^{-2}$ $8.8\times10^{-2}$ $8.8\times10^{-3}$ $2.6\times10^{-2}$ $2.6\times10^{-2}$ $2.1\times10^{-2}$ $1.7\times10^{-2}$ $2.2\times10^{-2}$ $2.4\times10^{-2}$ | 6700 6500 6600 4300 5000 | Brockbank (2013) Plyasunov et al. (2004) Ammari and Schroen (2019) Meynier et al. (2003) Kaneko et al. (1994) Mackay et al. (2006c) Mackay et al. (1995) Meylan and Howard (1991) Hine and Mookerjee (1975) Yaws (2003) Goldstein (1982) | L L M M M V V V V X X | 1 11 38 14 237 298 |



Table A3.8: Esters (RCOOR) (...continued)

| Substance Formula (Trivial Name) [CAS Registry Number] InChIKey | $H_s^{cp}$ (at $T^\ominus$) $\left[\dfrac{\mathrm{mol}}{\mathrm{m^3\,Pa}}\right]$ | $\dfrac{\mathrm{d}\ln H_s^{cp}}{\mathrm{d}(1/T)}$ [K] | Reference | Type | Note |
|---|---|---|---|---|---|
| | $1.7\times10^{-2}$ | | Meynier et al. (2003) | C | |
| | $5.3\times10^{-2}$ | | Keshavarz et al. (2022) | Q | |
| | $7.0$ | | Abney (2021) | Q | 399 |
| | $3.3\times10^{-2}$ | | Duchowicz et al. (2020) | Q | 299 |
| | $1.2\times10^{-2}$ | | Gharagheizi et al. (2012) | Q | |
| | $2.5\times10^{-2}$ | | Raventos-Duran et al. (2010) | Q | 271, 243 |
| | $2.5\times10^{-2}$ | | Raventos-Duran et al. (2010) | Q | 244 |
| | $2.0\times10^{-2}$ | | Raventos-Duran et al. (2010) | Q | 245 |
| | $2.7\times10^{-2}$ | | Gharagheizi et al. (2010) | Q | 246 |
| | $2.6\times10^{-2}$ | | Hilal et al. (2008) | Q | |
| | $3.2\times10^{-2}$ | | Modarresi et al. (2007) | Q | 67 |
| | $1.8\times10^{-2}$ | | Yaffe et al. (2003) | Q | 248, 249 |
| | $1.6\times10^{-2}$ | | Yao et al. (2002) | Q | 229 |
| | $2.4\times10^{-2}$ | | English and Carroll (2001) | Q | 230, 231 |
| | $3.8\times10^{-2}$ | | Katritzky et al. (1998) | Q | |
| | $1.7\times10^{-2}$ | | Suzuki et al. (1992) | Q | 232 |
| | $1.8\times10^{-2}$ | | Meylan and Howard (1991) | Q | |
| | $1.8\times10^{-2}$ | | Nirmalakhandan and Speece (1988) | Q | |
| | $1.7\times10^{-2}$ | | Duchowicz et al. (2020) | ? | 185, 21 |
| | $2.1\times10^{-2}$ | | Yaws (1999) | ? | 21 |
| | $1.7\times10^{-2}$ | | Abraham et al. (1990) | ? | |
| hexyl ethanoate $CH_3COOC_6H_{13}$ (hexyl acetate) [142-92-7] AOGQPLXWSUTHQB-UHFFFAOYSA-N | $2.0\times10^{-2}$ | 7100 | Brockbank (2013) | L | 1, 505 |
| | $1.7\times10^{-2}$ | 7300 | Plyasunov et al. (2004) | L | |
| | $1.4\times10^{-2}$ | | Souchon et al. (2004) | M | |
| | $1.5\times10^{-2}$ | | Karl et al. (2003) | M | |
| | $5.2\times10^{-3}$ | | Mackay et al. (2006c) | V | |
| | $5.2\times10^{-3}$ | | Mackay et al. (1995) | V | |
| | $1.8\times10^{-2}$ | | Hine and Mookerjee (1975) | V | |
| | $2.0\times10^{-2}$ | | Yaws (2003) | X | 258 |
| | $2.0\times10^{-2}$ | | Yaws (2003) | X | 237 |
| | $3.6\times10^{-2}$ | | Dupeux et al. (2022) | Q | 259 |
| | $7.1\times10^{-2}$ | | Keshavarz et al. (2022) | Q | |
| | $8.9\times10^{-2}$ | | Duchowicz et al. (2020) | Q | 184 |
| | $1.1\times10^{-2}$ | | Gharagheizi et al. (2012) | Q | |
| | $1.7\times10^{-2}$ | | Gharagheizi et al. (2010) | Q | 246 |
| | $1.4\times10^{-2}$ | | Hilal et al. (2008) | Q | |
| | $2.2\times10^{-2}$ | | Modarresi et al. (2007) | Q | 67 |
| | $2.0\times10^{-2}$ | | Yaffe et al. (2003) | Q | 248, 249 |
| | $2.0\times10^{-2}$ | | Yao et al. (2002) | Q | 229, 267 |
| | $1.7\times10^{-2}$ | | English and Carroll (2001) | Q | 230, 231 |
| | $3.6\times10^{-2}$ | | Katritzky et al. (1998) | Q | |
| | $9.9\times10^{-3}$ | | Russell et al. (1992) | Q | 279 |
| | $1.5\times10^{-2}$ | | Suzuki et al. (1992) | Q | 232 |
| | $2.2\times10^{-2}$ | | Nirmalakhandan and Speece (1988) | Q | |
| | $1.9\times10^{-2}$ | | Duchowicz et al. (2020) | ? | 185, 21 |
| | $2.0\times10^{-2}$ | | Yaws (1999) | ? | 21 |
| | $1.8\times10^{-2}$ | | Abraham et al. (1990) | ? | |



Table A3.8: Esters (RCOOR) (...continued)

| Substance<br>Formula<br>(Trivial Name)<br>[CAS Registry Number]<br>InChIKey | $H_s^{cp}$<br>(at $T^{\ominus}$)<br>$\left[\dfrac{\text{mol}}{\text{m}^3\,\text{Pa}}\right]$ | $\dfrac{\mathrm{d}\ln H_s^{cp}}{\mathrm{d}(1/T)}$<br><br>[K] | Reference | Type | Note |
|---|---|---|---|---|---|
| 4-methyl-2-pentyl ethanoate<br>$C_8H_{16}O_2$<br>(4-methyl-2-pentyl acetate)<br>[108-84-9]<br>CPIVYSAVIPTCCX-UHFFFAOYSA-N | $1.7\times10^{-2}$<br>$2.0\times10^{-2}$<br>$1.2\times10^{-2}$<br>$1.2\times10^{-2}$<br>$1.1\times10^{-2}$<br>$2.2\times10^{-2}$<br>$1.8\times10^{-2}$<br>$3.4\times10^{-2}$ | | HSDB (2015)<br>Raventos-Duran et al. (2010)<br>Raventos-Duran et al. (2010)<br>Raventos-Duran et al. (2010)<br>Hilal et al. (2008)<br>Modarresi et al. (2007)<br>Yaffe et al. (2003)<br>Katritzky et al. (1998) | V<br>Q<br>Q<br>Q<br>Q<br>Q<br>Q<br>Q | <br>242, 243<br>244<br>245<br><br>67<br>248, 249<br> |
| cyclohexyl ethanoate<br>$C_8H_{14}O_2$<br>(cyclohexyl acetate)<br>[622-45-7]<br>YYLLIJHXUHJATK-UHFFFAOYSA-N | $1.0\times10^{-1}$<br>$8.2\times10^{-2}$ | 7000 | Brockbank (2013)<br>HSDB (2015) | L<br>Q | 1, 506<br>99 |
| ethanol, 2-(2-ethoxyethoxy)-, acetate<br>$C_8H_{16}O_4$<br>(diethylene glycol monoethyl ether acetate)<br>[112-15-2]<br>FPZWZCWUIYYYBU-UHFFFAOYSA-N | $4.3\times10^2$ | | HSDB (2015) | V | |
| phenyl ethanoate<br>$C_8H_8O_2$<br>(phenyl acetate)<br>[122-79-2]<br>IPBVNPXQWQGGJP-UHFFFAOYSA-N | 1.3<br>$1.5\times10^{-1}$ | 7200 | Brockbank (2013)<br>HSDB (2015) | L<br>Q | 1<br>99 |
| heptyl ethanoate<br>$C_9H_{18}O_2$<br>(heptyl acetate)<br>[112-06-1]<br>ZCZSIDMEHXZRLG-UHFFFAOYSA-N | $1.2\times10^{-2}$<br>$1.1\times10^{-2}$<br>$2.4\times10^{-2}$<br>$1.1\times10^{-2}$<br>$1.3\times10^{-2}$ | | Yaws (2003)<br>Yaws (2003)<br>Dupeux et al. (2022)<br>Gharagheizi et al. (2012)<br>Gharagheizi et al. (2010)<br>Brockbank (2013) | X<br>X<br>Q<br>Q<br>Q<br>W | 258<br>237<br>259<br><br>246<br>507 |
| phenylmethyl ethanoate<br>$C_9H_{10}O_2$<br>(benzyl acetate)<br>[140-11-4]<br>QUKGYYKBILRGFE-UHFFFAOYSA-N | $8.5\times10^{-1}$<br>$8.7\times10^{-1}$<br>$9.0\times10^{-1}$<br>$7.6\times10^{-1}$<br>$5.2\times10^{-1}$<br>$7.0\times10^{-1}$<br>$9.2\times10^{-1}$<br>$5.4\times10^{-1}$ | 6800 | Brockbank (2013)<br>Duchowicz et al. (2020)<br>HSDB (2015)<br>Dupeux et al. (2022)<br>Duchowicz et al. (2020)<br>Modarresi et al. (2007)<br>Yaffe et al. (2003)<br>Katritzky et al. (1998) | L<br>V<br>V<br>Q<br>Q<br>Q<br>Q<br>Q | 1<br>186<br><br>259<br><br>67<br>248, 249<br> |



Table A3.8: Esters (RCOOR) (...continued)

| Substance Formula (Trivial Name) [CAS Registry Number] InChIKey | $H_s^{cp}$ (at $T^{\ominus}$) $\left[\dfrac{\text{mol}}{\text{m}^3\,\text{Pa}}\right]$ | $\dfrac{\text{d}\ln H_s^{cp}}{\text{d}(1/T)}$ [K] | Reference | Type | Note |
|---|---|---|---|---|---|
| octyl ethanoate $CH_3COOC_8H_{17}$ (octyl acetate) [112-14-1] YLYBTZIQSIBWLI-UHFFFAOYSA-N | $8.6\times10^{-3}$ $8.6\times10^{-3}$ $1.8\times10^{-2}$ $1.2\times10^{-2}$ $1.0\times10^{-2}$ | | Yaws (2003) Yaws (2003) Dupeux et al. (2022) Gharagheizi et al. (2012) Gharagheizi et al. (2010) Brockbank (2013) | X X Q Q Q W | 258 237 259 246 508 |
| 2-ethylhexyl ethanoate $C_{10}H_{20}O_2$ (2-ethylhexyl acetate) [103-09-3] WOYWLLHHWAMFCB-UHFFFAOYSA-N | $1.1\times10^{-2}$ $1.1\times10^{-2}$ $1.3\times10^{-2}$ $6.6\times10^{-3}$ $2.6\times10^{-3}$ $1.0\times10^{-2}$ $9.8\times10^{-3}$ $1.3\times10^{-2}$ | | Mackay et al. (2006c) Mackay et al. (1995) Yaws (2003) HSDB (2015) Gharagheizi et al. (2012) Gharagheizi et al. (2010) Yao et al. (2002) Yaws (1999) | V V X Q Q Q Q ? | 237 99 246 229 21 |
| nonyl ethanoate $C_{11}H_{22}O_2$ (nonyl acetate) [143-13-5] GJQIMXVRFNLMTB-UHFFFAOYSA-N | $4.1\times10^{-2}$ $2.1\times10^{-2}$ | | Yaws (2003) Dupeux et al. (2022) | X Q | 258 259 |
| decyl ethanoate $C_{12}H_{24}O_2$ (decyl acetate) [112-17-4] NUPSHWCALHZGOV-UHFFFAOYSA-N | $2.8\times10^{-3}$ $1.2\times10^{-2}$ $8.0\times10^{-3}$ | | Yaws (2003) Dupeux et al. (2022) Gharagheizi et al. (2012) | X Q Q | 258 259 |
| undecyl ethanoate $C_{13}H_{26}O_2$ (undecyl acetate) [1731-81-3] CKQGCFFDQIFZFA-UHFFFAOYSA-N | $1.5\times10^{-2}$ | | Gharagheizi et al. (2012) | Q | |
| ethanol, 2,2'-[1,2-ethanediylbis(oxy)]bis-, diacetate $C_{10}H_{18}O_6$ (triethylene glycol, diacetate) [111-21-7] OVOUKWFJRHALDD-UHFFFAOYSA-N | $3.7\times10^{7}$ | | HSDB (2015) | Q | 99 |
| 1-methoxy-2-propyl ethanoate $C_6H_{12}O_3$ (1-methoxy-2-propyl acetate) [108-65-6] LLHKCFNBLRBOGN-UHFFFAOYSA-N | 2.9 $8.7\times10^{-1}$ 2.5 1.6 2.5 $9.9\times10^{-1}$ $6.3\times10^{-1}$ | | Duchowicz et al. (2020) Duchowicz et al. (2020) Raventos-Duran et al. (2010) Raventos-Duran et al. (2010) Raventos-Duran et al. (2010) Hilal et al. (2008) Modarresi et al. (2007) | V Q Q Q Q Q Q | 186 242, 243 244 245 67 |



Table A3.8: Esters (RCOOR) (...continued)

| Substance<br>Formula<br>(Trivial Name)<br>[CAS Registry Number]<br>InChIKey | $H_s^{cp}$<br>(at $T^{\ominus}$)<br>$\left[\dfrac{\text{mol}}{\text{m}^3\,\text{Pa}}\right]$ | $\dfrac{\text{d}\ln H_s^{cp}}{\text{d}(1/T)}$<br><br>[K] | Reference | Type | Note |
|---|---|---|---|---|---|
| 2-ethoxyethyl ethanoate | 3.0 | | Brockbank (2013) | L | |
| $C_6H_{12}O_3$ | 3.0 | | Hovorka et al. (2002) | M | 38 |
| (2-ethoxyethyl acetate) | 1.5 | | Johanson and Dynésius (1988) | M | 14 |
| [111-15-9] | 3.4 | | Keshavarz et al. (2022) | Q | |
| SVONRAPFKPVNKG-UHFFFAOYSA-N | $7.5\times10^{-1}$ | | Duchowicz et al. (2020) | Q | 184 |
| | 2.5 | | Raventos-Duran et al. (2010) | Q | 242, 243 |
| | 2.5 | | Raventos-Duran et al. (2010) | Q | 244 |
| | 2.5 | | Raventos-Duran et al. (2010) | Q | 245 |
| | 1.9 | | Hilal et al. (2008) | Q | |
| | $7.2\times10^{-1}$ | | Modarresi et al. (2007) | Q | 67 |
| | 3.1 | | Duchowicz et al. (2020) | ? | 185, 21 |
| | 7.0 | | Yaws (1999) | ? | 21, 12 |
| 2-butoxyethyl ethanoate | 1.8 | | Brockbank (2013) | L | |
| $C_8H_{16}O_3$ | 1.8 | 25000 | Kim et al. (2000) | M | |
| (butyl cellosolve acetate) | 6.1 | | Keshavarz et al. (2022) | Q | |
| [112-07-2] | $9.6\times10^{-1}$ | | Duchowicz et al. (2020) | Q | 299 |
| NQBXSWAWVZHKBZ-UHFFFAOYSA-N | 1.2 | | Raventos-Duran et al. (2010) | Q | 242, 243 |
| | 1.6 | | Raventos-Duran et al. (2010) | Q | 244 |
| | 1.6 | | Raventos-Duran et al. (2010) | Q | 245 |
| | 1.3 | | Hilal et al. (2008) | Q | |
| | $5.2\times10^{-1}$ | | Modarresi et al. (2007) | Q | 67 |
| | 1.8 | | Duchowicz et al. (2020) | ? | 185, 21 |
| 2-(2-butoxyethoxy)-ethanol, ethanoate | $2.8\times10^1$ | | Duchowicz et al. (2020) | V | 186 |
| $C_{10}H_{20}O_4$ | $2.8\times10^1$ | | HSDB (2015) | V | |
| [124-17-4] | $1.1\times10^1$ | | Duchowicz et al. (2020) | Q | |
| VXQBJTKSVGFQOL-UHFFFAOYSA-N | $6.2\times10^1$ | | Raventos-Duran et al. (2010) | Q | 242, 243 |
| | $7.8\times10^1$ | | Raventos-Duran et al. (2010) | Q | 244 |
| | $9.9\times10^1$ | | Raventos-Duran et al. (2010) | Q | 245 |
| | $4.1\times10^1$ | | Hilal et al. (2008) | Q | |
| | 7.2 | | Modarresi et al. (2007) | Q | 67 |
| 1,2-ethanediol, diethanoate | $1.2\times10^2$ | | Duchowicz et al. (2020) | V | 186 |
| $C_6H_{10}O_4$ | $1.2\times10^2$ | | HSDB (2015) | V | |
| [111-55-7] | $1.3\times10^2$ | | Yaws (2003) | X | 237, 87 |
| JTXMVXSTHSMVQF-UHFFFAOYSA-N | 6.4 | | Duchowicz et al. (2020) | Q | |
| | $1.6\times10^1$ | | Raventos-Duran et al. (2010) | Q | 242, 243 |
| | $1.6\times10^1$ | | Raventos-Duran et al. (2010) | Q | 244 |
| | $2.0\times10^1$ | | Raventos-Duran et al. (2010) | Q | 245 |
| | $1.4\times10^2$ | | Gharagheizi et al. (2010) | Q | 246 |
| | $1.3\times10^1$ | | Hilal et al. (2008) | Q | |
| | $9.3\times10^1$ | | Yaws (1999) | ? | 21, 87 |
| geranyl acetate | $4.1\times10^{-3}$ | | HSDB (2015) | Q | 99 |
| $C_{12}H_{20}O_2$ | | | | | |
| [105-87-3] | | | | | |
| HIGQPQRQIQDZMP-DHZHZOJOSA-N | | | | | |





Table A3.8: Esters (RCOOR) (... continued)

| Substance Formula (Trivial Name) [CAS Registry Number] InChIKey | $H_s^{cp}$ (at $T^{\ominus}$) $\left[\dfrac{\text{mol}}{\text{m}^3\,\text{Pa}}\right]$ | $\dfrac{\text{d}\ln H_s^{cp}}{\text{d}(1/T)}$ [K] | Reference | Type | Note |
|---|---|---|---|---|---|
| linalyl acetate $C_{12}H_{20}O_2$ [115-95-7] UWKAYLJWKGQEPM-UHFFFAOYSA-N | $1.8\times10^{-2}$ $5.8\times10^{-3}$ | | Dupeux et al. (2022) HSDB (2015) | Q Q | 259 99 |
| cyclohexanol, 5-methyl-2-(1-methylethyl)-, acetate $C_{12}H_{22}O_2$ (menthyl acetate) [16409-45-3] XHXUANMFYXWVNG-UHFFFAOYSA-N | $1.2\times10^{-2}$ | | HSDB (2015) | Q | 99 |
| methyl propanoate $C_2H_5COOCH_3$ (methyl propionate) [554-12-1] RJUFJBKOKNCXHH-UHFFFAOYSA-N | $5.8\times10^{-2}$ $5.5\times10^{-2}$ $5.2\times10^{-2}$ $5.7\times10^{-2}$ $6.1\times10^{-2}$ $6.1\times10^{-2}$ $1.3\times10^{-1}$ $6.1\times10^{-2}$ $5.4\times10^{-2}$ $2.2\times10^{-2}$ $2.0\times10^{-1}$ $6.2\times10^{-2}$ $3.9\times10^{-2}$ $3.9\times10^{-2}$ $3.9\times10^{-2}$ $5.2\times10^{-2}$ $5.8\times10^{-2}$ $5.3\times10^{-2}$ $5.3\times10^{-2}$ $3.9\times10^{-2}$ $4.4\times10^{-2}$ $4.0\times10^{-2}$ $5.7\times10^{-2}$ $5.2\times10^{-2}$ $5.8\times10^{-2}$ $5.7\times10^{-2}$ | 5700 5400 3900 5400 5000 | Brockbank (2013) Plyasunov et al. (2004) Ioffe et al. (1984) Buttery et al. (1969) Mackay et al. (2006c) Mackay et al. (1995) Djerki and Laub (1988) Hine and Mookerjee (1975) Bagno et al. (1991) Keshavarz et al. (2022) Duchowicz et al. (2020) Raventos-Duran et al. (2010) Raventos-Duran et al. (2010) Raventos-Duran et al. (2010) Hilal et al. (2008) Modarresi et al. (2007) Yaffe et al. (2003) Yao et al. (2002) English and Carroll (2001) Katritzky et al. (1998) Suzuki et al. (1992) Nirmalakhandan and Speece (1988) Duchowicz et al. (2020) Yaws (1999) Betterton (1992) Abraham et al. (1990) | L L M M V V V V T Q Q Q Q Q Q Q Q Q Q Q Q Q ? ? ? ? | 1 473 184 242, 243 244 245 67 248, 249 229 230, 231 232 185, 21 21 509 |
| methyl 2-hydroxypropanoate $C_4H_8O_3$ (methyl lactate) [547-64-8] LPEKGGXMPWTOCB-UHFFFAOYSA-N | 9.1 4.7 $1.1\times10^2$ $1.4\times10^1$ $1.2\times10^3$ | 5000 | Sanz and Gmehling (2005) Wang et al. (2017) Wang et al. (2017) Wang et al. (2017) HSDB (2015) | M Q Q Q Q | 80, 238 80, 239 80, 240 99 |





Table A3.8: Esters (RCOOR) (...continued)

| Substance Formula (Trivial Name) [CAS Registry Number] InChIKey | $H_s^{cp}$ (at $T^{\ominus}$) $\left[\dfrac{\text{mol}}{\text{m}^3\,\text{Pa}}\right]$ | $\dfrac{\text{d}\ln H_s^{cp}}{\text{d}(1/T)}$ [K] | Reference | Type | Note |
|---|---|---|---|---|---|
| ethyl propanoate | $3.8\times10^{-2}$ | 5900 | Brockbank (2013) | L | 1 |
| $C_2H_5COOC_2H_5$ | $4.1\times10^{-2}$ | 6000 | Plyasunov et al. (2004) | L | |
| (ethyl propionate) | $4.1\times10^{-2}$ | 5900 | Fenclová et al. (2014) | M | 1 |
| [105-37-3] | $3.9\times10^{-2}$ | | Duchowicz et al. (2020) | V | 186 |
| FKRCODPIKNYEAC-UHFFFAOYSA-N | $3.9\times10^{-2}$ | | HSDB (2015) | V | |
| | $3.8\times10^{-2}$ | | Mackay et al. (2006c) | V | |
| | $3.8\times10^{-2}$ | | Mackay et al. (1995) | V | |
| | $3.7\times10^{-2}$ | | Abraham (1984) | V | |
| | $4.5\times10^{-2}$ | | Hine and Mookerjee (1975) | V | |
| | $4.5\times10^{-2}$ | | Yaws (2003) | X | 258 |
| | $4.5\times10^{-2}$ | | Yaws (2003) | X | 237 |
| | $4.1\times10^{-2}$ | | Dupeux et al. (2022) | Q | 259 |
| | $7.7\times10^{-2}$ | | Duchowicz et al. (2020) | Q | |
| | $4.5\times10^{-2}$ | | Li et al. (2014) | Q | 241 |
| | $2.2\times10^{-2}$ | | Gharagheizi et al. (2012) | Q | |
| | $3.9\times10^{-2}$ | | Raventos-Duran et al. (2010) | Q | 271, 243 |
| | $2.5\times10^{-2}$ | | Raventos-Duran et al. (2010) | Q | 244 |
| | $3.1\times10^{-2}$ | | Raventos-Duran et al. (2010) | Q | 245 |
| | $4.7\times10^{-2}$ | | Gharagheizi et al. (2010) | Q | 246 |
| | $2.6\times10^{-2}$ | | Hilal et al. (2008) | Q | |
| | $4.4\times10^{-2}$ | | Modarresi et al. (2007) | Q | 67 |
| | $4.6\times10^{-2}$ | | Yaffe et al. (2003) | Q | 248, 249 |
| | $3.5\times10^{-2}$ | | Yao et al. (2002) | Q | 229 |
| | $4.0\times10^{-2}$ | | English and Carroll (2001) | Q | 230, 231 |
| | $3.9\times10^{-2}$ | | Katritzky et al. (1998) | Q | |
| | $3.4\times10^{-2}$ | | Suzuki et al. (1992) | Q | 232 |
| | $3.5\times10^{-2}$ | | Nirmalakhandan and Speece (1988) | Q | |
| | $4.5\times10^{-2}$ | | Yaws (1999) | ? | 21 |
| | $3.8\times10^{-2}$ | | Abraham et al. (1990) | ? | |
| propyl propanoate | $2.6\times10^{-2}$ | 6400 | Brockbank (2013) | L | 1, 510 |
| $C_2H_5COOC_3H_7$ | $2.9\times10^{-2}$ | 6200 | Plyasunov et al. (2004) | L | |
| (propyl propionate) | $2.5\times10^{-2}$ | | Duchowicz et al. (2020) | V | 186 |
| [106-36-5] | $2.5\times10^{-2}$ | | Abraham (1984) | V | |
| MCSINKKTEDDPNK-UHFFFAOYSA-N | $2.5\times10^{-2}$ | | Hine and Mookerjee (1975) | V | |
| | $8.2\times10^{-2}$ | | Duchowicz et al. (2020) | Q | |
| | $3.1\times10^{-2}$ | | Raventos-Duran et al. (2010) | Q | 271, 243 |
| | $2.0\times10^{-2}$ | | Raventos-Duran et al. (2010) | Q | 244 |
| | $2.5\times10^{-2}$ | | Raventos-Duran et al. (2010) | Q | 245 |
| | $2.0\times10^{-2}$ | | Hilal et al. (2008) | Q | |
| | $3.6\times10^{-2}$ | | Modarresi et al. (2007) | Q | 67 |
| | $2.7\times10^{-2}$ | | Yaffe et al. (2003) | Q | 248, 249 |
| | $2.6\times10^{-2}$ | | Yao et al. (2002) | Q | 229 |
| | $3.1\times10^{-2}$ | | English and Carroll (2001) | Q | 230, 274 |
| | $3.9\times10^{-2}$ | | Katritzky et al. (1998) | Q | |
| | $2.6\times10^{-2}$ | | Suzuki et al. (1992) | Q | 232 |
| | $2.8\times10^{-2}$ | | Nirmalakhandan and Speece (1988) | Q | |
| | $2.1\times10^{-2}$ | | Yaws (1999) | ? | 21 |



Table A3.8: Esters (RCOOR) (...continued)

| Substance Formula (Trivial Name) [CAS Registry Number] InChIKey | $H_s^{cp}$ (at $T^{\ominus}$) $\left[\dfrac{\mathrm{mol}}{\mathrm{m^3\,Pa}}\right]$ | $\dfrac{\mathrm{d}\ln H_s^{cp}}{\mathrm{d}(1/T)}$ [K] | Reference | Type | Note |
|---|---|---|---|---|---|
| | $2.5\times10^{-2}$ | | Abraham et al. (1990) | ? | |
| isopropyl propanoate $C_2H_5COOC_3H_7$ (isopropyl propionate) [637-78-5] IJMWOMHMDSDKGK-UHFFFAOYSA-N | $1.8\times10^{-2}$ | | Plyasunov et al. (2004) | L | |
| | $1.7\times10^{-2}$ | | Duchowicz et al. (2020) | V | 186 |
| | $1.7\times10^{-2}$ | | Meylan and Howard (1991) | V | |
| | $1.7\times10^{-2}$ | | Hine and Mookerjee (1975) | V | |
| | $3.2\times10^{-2}$ | | Duchowicz et al. (2020) | Q | |
| | $1.7\times10^{-2}$ | | Hilal et al. (2008) | Q | |
| | $3.1\times10^{-2}$ | | Modarresi et al. (2007) | Q | 67 |
| | $2.3\times10^{-2}$ | | Suzuki et al. (1992) | Q | 232 |
| | $2.4\times10^{-2}$ | | Meylan and Howard (1991) | Q | |
| | $2.5\times10^{-2}$ | | Nirmalakhandan and Speece (1988) | Q | |
| | $1.7\times10^{-2}$ | | Abraham et al. (1990) | ? | |
| butyl propanoate $C_7H_{14}O_2$ (butyl propionate) [590-01-2] BTMVHUNTONAYDX-UHFFFAOYSA-N | $2.2\times10^{-2}$ | 6900 | Brockbank (2013) | L | 1 |
| | $2.0\times10^{-2}$ | 7000 | Plyasunov et al. (2004) | L | |
| | $2.0\times10^{-2}$ | | Duchowicz et al. (2020) | V | 186 |
| | $3.3\times10^{-2}$ | | Yaws (2003) | X | 258 |
| | $3.3\times10^{-2}$ | | Yaws (2003) | X | 237, 12 |
| | $2.6\times10^{-2}$ | | Dupeux et al. (2022) | Q | 259 |
| | $8.6\times10^{-2}$ | | Duchowicz et al. (2020) | Q | |
| | $8.9\times10^{-3}$ | | Gharagheizi et al. (2012) | Q | |
| | $2.7\times10^{-2}$ | | Gharagheizi et al. (2010) | Q | 246 |
| | $3.2\times10^{-2}$ | | Modarresi et al. (2007) | Q | 67 |
| | $6.2\times10^{-3}$ | | Yaffe et al. (2003) | Q | 248, 272 |
| | $2.3\times10^{-2}$ | | Yao et al. (2002) | Q | 229 |
| | $3.7\times10^{-2}$ | | Katritzky et al. (1998) | Q | |
| | $1.4\times10^{-2}$ | | Yaws (1999) | ? | 21 |
| (2-methylpropyl)-propanoate $C_7H_{14}O_2$ (isobutyl propionate) [540-42-1] FZXRXKLUIMKDEL-UHFFFAOYSA-N | $1.5\times10^{-2}$ | 6600 | Plyasunov et al. (2004) | L | |
| | $1.5\times10^{-2}$ | | Duchowicz et al. (2020) | V | 186 |
| | $3.3\times10^{-2}$ | | Duchowicz et al. (2020) | Q | |
| | $1.8\times10^{-2}$ | | Hilal et al. (2008) | Q | |
| | | 5900 | Kühne et al. (2005) | Q | |
| | $6.2\times10^{-3}$ | | Yaffe et al. (2003) | Q | 248, 249 |
| | $1.9\times10^{-2}$ | | Nirmalakhandan et al. (1997) | Q | |
| | | 7300 | Kühne et al. (2005) | ? | |
| *tert*-butyl propanoate $C_7H_{14}O_2$ [20487-40-5] JAELLLITIZHOGQ-UHFFFAOYSA-N | $7.5\times10^{-3}$ | | Ebert et al. (2023) | ? | 365 |
| pentyl propanoate $C_2H_5COOC_5H_{11}$ (amyl propionate) [624-54-4] TWSRVQVEYJNFKQ-UHFFFAOYSA-N | $1.9\times10^{-2}$ | | Plyasunov et al. (2004) | L | |
| | $1.2\times10^{-2}$ | | Duchowicz et al. (2020) | V | 186 |
| | $1.4\times10^{-2}$ | | Abraham (1984) | V | |
| | $1.2\times10^{-2}$ | | Hine and Mookerjee (1975) | V | |
| | $8.9\times10^{-2}$ | | Duchowicz et al. (2020) | Q | |
| | $2.0\times10^{-2}$ | | Raventos-Duran et al. (2010) | Q | 242, 243 |
| | $1.2\times10^{-2}$ | | Raventos-Duran et al. (2010) | Q | 244 |



Table A3.8: Esters (RCOOR) (...continued)

| Substance<br>Formula<br>(Trivial Name)<br>[CAS Registry Number]<br>InChIKey | $H_s^{cp}$<br>(at $T^{\ominus}$)<br>$\left[\dfrac{\text{mol}}{\text{m}^3\,\text{Pa}}\right]$ | $\dfrac{\text{d}\ln H_s^{cp}}{\text{d}(1/T)}$<br><br>[K] | Reference | Type | Note |
|---|---|---|---|---|---|
| | $1.2\times10^{-2}$ | | Raventos-Duran et al. (2010) | Q | 245 |
| | $1.2\times10^{-2}$ | | Hilal et al. (2008) | Q | |
| | $2.6\times10^{-2}$ | | Modarresi et al. (2007) | Q | 67 |
| | $1.2\times10^{-2}$ | | Yaffe et al. (2003) | Q | 248, 249 |
| | $1.7\times10^{-2}$ | | English and Carroll (2001) | Q | 230, 231 |
| | $3.4\times10^{-2}$ | | Katritzky et al. (1998) | Q | |
| | $2.2\times10^{-2}$ | | Nirmalakhandan et al. (1997) | Q | |
| | $1.5\times10^{-2}$ | | Suzuki et al. (1992) | Q | 232 |
| | $1.8\times10^{-2}$ | | Nirmalakhandan and Speece (1988) | Q | |
| | $1.4\times10^{-2}$ | | Abraham et al. (1990) | ? | |
| propanoic acid, 2-hydroxy-, ethyl ester<br>$C_5H_{10}O_3$<br>(ethyl lactate)<br>[97-64-3]<br>LZCLXQDLBQLTDK-UHFFFAOYSA-N | $1.7\times10^{1}$<br><br>$1.7\times10^{1}$<br>3.1 | | Duchowicz et al. (2020)<br><br>HSDB (2015)<br>Duchowicz et al. (2020) | V<br><br>V<br>Q | 186 |
| ethyl 3-ethoxypropanoate<br>$C_7H_{14}O_3$<br>(ethyl 3-ethoxypropionate)<br>[763-69-9]<br>BHXIWUJLHYHGSJ-UHFFFAOYSA-N | $1.5\times10^{-2}$ | | Yaws (1999) | ? | 21 |
| propanoic acid, 2-phenylethyl ester<br>$C_{11}H_{14}O_2$<br>[122-70-3]<br>HVGZQCSMLUDISR-UHFFFAOYSA-N | $3.9\times10^{-1}$ | | HSDB (2015) | Q | 99 |
| methyl butanoate<br>$C_3H_7COOCH_3$<br>(methyl butyrate)<br>[623-42-7]<br>UUIQMZJEGPQKFD-UHFFFAOYSA-N | $3.9\times10^{-2}$<br>$4.2\times10^{-2}$<br>$3.7\times10^{-2}$<br>$3.6\times10^{-2}$<br>$4.8\times10^{-2}$<br>$3.6\times10^{-1}$<br>$3.7\times10^{-2}$<br><br>$3.5\times10^{-2}$<br>$3.6\times10^{-2}$<br>$4.4\times10^{-2}$<br>$2.9\times10^{-2}$<br>$2.3\times10^{-1}$<br>$4.8\times10^{-2}$<br>$1.2\times10^{-2}$<br>$3.9\times10^{-2}$<br>$2.5\times10^{-2}$<br>$3.1\times10^{-2}$<br>$2.8\times10^{-2}$<br>$4.0\times10^{-2}$ | 5700<br>5700<br><br><br><br>4400<br><br>5800 | Brockbank (2013)<br>Plyasunov et al. (2004)<br>Aprea et al. (2007)<br>Ioffe et al. (1984)<br>Buttery et al. (1969)<br>Djerki and Laub (1988)<br>Amoore and Buttery (1978)<br>Della Gatta et al. (1981)<br>Yaws (2003)<br>Nahon et al. (2000)<br>Dupeux et al. (2022)<br>Keshavarz et al. (2022)<br>Duchowicz et al. (2020)<br>Li et al. (2014)<br>Gharagheizi et al. (2012)<br>Raventos-Duran et al. (2010)<br>Raventos-Duran et al. (2010)<br>Raventos-Duran et al. (2010)<br>Hilal et al. (2008)<br>Modarresi et al. (2007) | L<br>L<br>M<br>M<br>M<br>V<br>V<br>T<br>X<br>C<br>Q<br>Q<br>Q<br>Q<br>Q<br>Q<br>Q<br>Q<br>Q<br>Q | 1, 511<br><br><br><br><br><br><br><br>258<br>14<br>259<br><br>299<br>241<br><br>242, 243<br>244<br>245<br><br>67 |



Table A3.8: Esters (RCOOR) (...continued)

| Substance Formula (Trivial Name) [CAS Registry Number] InChIKey | $H_s^{cp}$ (at $T^\ominus$) $\left[\dfrac{\text{mol}}{\text{m}^3\,\text{Pa}}\right]$ | $\dfrac{\text{d}\ln H_s^{cp}}{\text{d}(1/T)}$ [K] | Reference | Type | Note |
|---|---|---|---|---|---|
| | $4.6\times10^{-2}$ | | Yaffe et al. (2003) | Q | 248, 272 |
| | $3.2\times10^{-2}$ | | Yao et al. (2002) | Q | 229 |
| | $4.0\times10^{-2}$ | | English and Carroll (2001) | Q | 230, 231 |
| | $3.9\times10^{-2}$ | | Katritzky et al. (1998) | Q | |
| | $1.5\times10^{-1}$ | | Russell et al. (1992) | Q | 279 |
| | $3.4\times10^{-2}$ | | Suzuki et al. (1992) | Q | 232 |
| | $3.2\times10^{-2}$ | | Nirmalakhandan and Speece (1988) | Q | |
| | $4.8\times10^{-2}$ | | Duchowicz et al. (2020) | ? | 185, 21 |
| | $3.5\times10^{-2}$ | | Yaws (1999) | ? | 21 |
| | $4.8\times10^{-2}$ | | Abraham et al. (1990) | ? | |
| ethyl butanoate $C_3H_7COOC_2H_5$ (ethyl butyrate) [105-54-4] OBNCKNCVKJNDBV-UHFFFAOYSA-N | $2.5\times10^{-2}$ | 6100 | Brockbank (2013) | L | 1 |
| | $2.9\times10^{-2}$ | 6300 | Plyasunov et al. (2004) | L | |
| | $2.9\times10^{-2}$ | 6400 | Fenclová et al. (2014) | M | 1 |
| | $2.4\times10^{-2}$ | | Aprea et al. (2007) | M | |
| | $2.6\times10^{-2}$ | | Souchon et al. (2004) | M | |
| | $2.1\times10^{-2}$ | | van Ruth et al. (2002) | M | 14 |
| | $2.1\times10^{-2}$ | | van Ruth and Villeneuve (2002) | M | 14, 361 |
| | $1.6\times10^{-2}$ | | van Ruth et al. (2001) | M | 14 |
| | $4.0\times10^{-2}$ | | Landy et al. (1996) | M | |
| | $2.4\times10^{-2}$ | | Landy et al. (1995) | M | |
| | $2.5\times10^{-2}$ | | HSDB (2015) | V | |
| | $2.4\times10^{-2}$ | | Mackay et al. (2006c) | V | |
| | $1.2\times10^{-2}$ | | Philippe et al. (2003) | V | 14 |
| | $2.4\times10^{-2}$ | | Mackay et al. (1995) | V | |
| | $2.8\times10^{-2}$ | | Abraham (1984) | V | |
| | $2.7\times10^{-2}$ | | Hine and Mookerjee (1975) | V | |
| | $2.8\times10^{-2}$ | | Yaws (2003) | X | 258 |
| | $2.8\times10^{-2}$ | | Yaws (2003) | X | 237, 87 |
| | $2.7\times10^{-2}$ | | Nahon et al. (2000) | C | 14 |
| | $4.5\times10^{-2}$ | | Dupeux et al. (2022) | Q | 259 |
| | 3.4 | | Abney (2021) | Q | 399 |
| | $2.4\times10^{-2}$ | | Savary et al. (2014) | Q | |
| | $1.6\times10^{-2}$ | | Gharagheizi et al. (2012) | Q | |
| | $3.1\times10^{-2}$ | | Raventos-Duran et al. (2010) | Q | 271, 243 |
| | $2.0\times10^{-2}$ | | Raventos-Duran et al. (2010) | Q | 244 |
| | $2.5\times10^{-2}$ | | Raventos-Duran et al. (2010) | Q | 245 |
| | $3.5\times10^{-2}$ | | Gharagheizi et al. (2010) | Q | 246 |
| | $2.0\times10^{-2}$ | | Hilal et al. (2008) | Q | |
| | $3.1\times10^{-2}$ | | Modarresi et al. (2007) | Q | 67 |
| | $2.7\times10^{-2}$ | | Yaffe et al. (2003) | Q | 248, 272 |
| | $2.8\times10^{-2}$ | | Yao et al. (2002) | Q | 229 |
| | $3.1\times10^{-2}$ | | English and Carroll (2001) | Q | 230, 231 |
| | $3.9\times10^{-2}$ | | Katritzky et al. (1998) | Q | |
| | $2.6\times10^{-2}$ | | Suzuki et al. (1992) | Q | 232 |
| | $2.8\times10^{-2}$ | | Nirmalakhandan and Speece (1988) | Q | |
| | $2.4\times10^{-2}$ | | Yaws (1999) | ? | 21, 87 |
| | $2.7\times10^{-2}$ | | Abraham et al. (1990) | ? | |



Table A3.8: Esters (RCOOR) (... continued)

| Substance Formula (Trivial Name) [CAS Registry Number] InChIKey | $H_s^{cp}$ (at $T^{\ominus}$) $\left[\dfrac{\mathrm{mol}}{\mathrm{m^3\,Pa}}\right]$ | $\dfrac{\mathrm{d}\ln H_s^{cp}}{\mathrm{d}(1/T)}$ [K] | Reference | Type | Note |
|---|---|---|---|---|---|
| propyl butanoate | $1.8\times10^{-2}$ | 6400 | Brockbank (2013) | L | 1 |
| $C_3H_7COOC_3H_7$ | $2.2\times10^{-2}$ | 6600 | Plyasunov et al. (2004) | L | |
| (propyl butyrate) | $1.6\times10^{-2}$ | | Duchowicz et al. (2020) | V | 186 |
| [105-66-8] | $1.6\times10^{-2}$ | | Meylan and Howard (1991) | V | |
| HUAZGNHGCJGYNP-UHFFFAOYSA-N | $1.9\times10^{-2}$ | | Hine and Mookerjee (1975) | V | |
| | $2.6\times10^{-2}$ | | Yaws (2003) | X | 237, 154 |
| | $8.6\times10^{-2}$ | | Duchowicz et al. (2020) | Q | |
| | $8.1\times10^{-3}$ | | Gharagheizi et al. (2012) | Q | |
| | $2.5\times10^{-2}$ | | Raventos-Duran et al. (2010) | Q | 242, 243 |
| | $1.2\times10^{-2}$ | | Raventos-Duran et al. (2010) | Q | 244 |
| | $2.0\times10^{-2}$ | | Raventos-Duran et al. (2010) | Q | 245 |
| | $2.7\times10^{-2}$ | | Gharagheizi et al. (2010) | Q | 246 |
| | $1.4\times10^{-2}$ | | Hilal et al. (2008) | Q | |
| | $2.8\times10^{-2}$ | | Modarresi et al. (2007) | Q | 67 |
| | $2.2\times10^{-2}$ | | Yao et al. (2002) | Q | 229 |
| | $2.3\times10^{-2}$ | | English and Carroll (2001) | Q | 230, 260 |
| | $3.6\times10^{-2}$ | | Katritzky et al. (1998) | Q | |
| | $2.0\times10^{-2}$ | | Suzuki et al. (1992) | Q | 232 |
| | $1.8\times10^{-2}$ | | Meylan and Howard (1991) | Q | |
| | $2.2\times10^{-2}$ | | Nirmalakhandan and Speece (1988) | Q | |
| | $1.6\times10^{-2}$ | | Yaws (1999) | ? | 21, 154 |
| | $1.9\times10^{-2}$ | | Abraham et al. (1990) | ? | |
| 2-propyl butanoate | $1.8\times10^{-2}$ | | Plyasunov et al. (2004) | L | |
| $C_7H_{14}O_2$ | $1.9\times10^{-2}$ | | Dupeux et al. (2022) | Q | 259 |
| (isopropyl butyrate) | | | | | |
| [638-11-9] | | | | | |
| FFOPEPMHKILNIT-UHFFFAOYSA-N | | | | | |
| butyl butanoate | $1.7\times10^{-2}$ | 7600 | Brockbank (2013) | L | 1 |
| $C_8H_{16}O_2$ | $1.6\times10^{-2}$ | 7600 | Plyasunov et al. (2004) | L | |
| (butyl butyrate) | $1.4\times10^{-2}$ | | Duchowicz et al. (2020) | V | 186 |
| [109-21-7] | $1.0\times10^{-2}$ | | Dupeux et al. (2022) | Q | 259 |
| XUPYJHCZDLZNFP-UHFFFAOYSA-N | $8.9\times10^{-2}$ | | Duchowicz et al. (2020) | Q | |
| | $2.0\times10^{-2}$ | | Raventos-Duran et al. (2010) | Q | 242, 243 |
| | $9.9\times10^{-3}$ | | Raventos-Duran et al. (2010) | Q | 244 |
| | $1.2\times10^{-2}$ | | Raventos-Duran et al. (2010) | Q | 245 |
| | $2.3\times10^{-2}$ | | Modarresi et al. (2007) | Q | 67 |
| | $3.4\times10^{-2}$ | | Katritzky et al. (1998) | Q | |
| | $1.1\times10^{-2}$ | | Yaws (1999) | ? | 21 |
| 2-methylpropyl butanoate | $1.3\times10^{-2}$ | | Hilal et al. (2008) | Q | |
| $C_8H_{16}O_2$ | | | | | |
| (2-methylpropyl butyrate) | | | | | |
| [539-90-2] | | | | | |
| RGFNRWTWDWVHDD-UHFFFAOYSA-N | | | | | |



Table A3.8: Esters (RCOOR) (. . . continued)

| Substance<br>Formula<br>(Trivial Name)<br>[CAS Registry Number]<br>InChIKey | $H_s^{cp}$<br>(at $T^{\ominus}$)<br>$\left[\dfrac{\mathrm{mol}}{\mathrm{m}^3\,\mathrm{Pa}}\right]$ | $\dfrac{\mathrm{d}\ln H_s^{cp}}{\mathrm{d}(1/T)}$<br><br>[K] | Reference | Type | Note |
|---|---|---|---|---|---|
| 2-methylpropyl<br>2-methylpropanoate | $7.4\times10^{-3}$ | 6900 | Brockbank (2013) | L | 1, 512 |
| C$_8$H$_{16}$O$_2$ | $7.8\times10^{-3}$ | 6700 | Plyasunov et al. (2004) | L | |
| (isobutyl isobutyrate) | $1.0\times10^{-2}$ | | Amoore and Buttery (1978) | M | |
| [97-85-8] | $1.2\times10^{-2}$ | | Duchowicz et al. (2020) | V | 186 |
| RXGUIWHIADMCFC-UHFFFAOYSA-N | $7.2\times10^{-3}$ | | Amoore and Buttery (1978) | V | |
| | $1.4\times10^{-2}$ | | Duchowicz et al. (2020) | Q | |
| | $1.8\times10^{-3}$ | | Gharagheizi et al. (2012) | Q | |
| | $2.0\times10^{-2}$ | | Raventos-Duran et al. (2010) | Q | 242, 243 |
| | $1.2\times10^{-2}$ | | Raventos-Duran et al. (2010) | Q | 244 |
| | $1.2\times10^{-2}$ | | Raventos-Duran et al. (2010) | Q | 245 |
| | $1.2\times10^{-2}$ | | Hilal et al. (2008) | Q | |
| | $1.0\times10^{-2}$ | | Modarresi et al. (2007) | Q | 67 |
| | $7.5\times10^{-3}$ | | Yaffe et al. (2003) | Q | 248, 249 |
| | $1.4\times10^{-2}$ | | English and Carroll (2001) | Q | 230, 231 |
| | $1.3\times10^{-2}$ | | Nirmalakhandan et al. (1997) | Q | |
| | $6.9\times10^{-3}$ | | Yaws (1999) | ? | 21 |
| | $7.0\times10^{-3}$ | | Abraham et al. (1990) | ? | |
| pentyl butanoate | $1.7\times10^{-2}$ | 6400 | Brockbank (2013) | L | 1 |
| C$_9$H$_{18}$O$_2$ | $9.8\times10^{-3}$ | | Dupeux et al. (2022) | Q | 259 |
| (pentyl butyrate) | | | | | |
| [540-18-1] | | | | | |
| CFNJLPHOBMVMNS-UHFFFAOYSA-N | | | | | |
| 3-methylbutyl butanoate | $1.1\times10^{-2}$ | 7500 | Brockbank (2013) | L | 1 |
| C$_9$H$_{18}$O$_2$ | | | | | |
| (isopentyl butyrate) | | | | | |
| [106-27-4] | | | | | |
| PQLMXFQTAMDXIZ-UHFFFAOYSA-N | | | | | |
| methyl 2-methylpropanoate | $2.6\times10^{-2}$ | 5600 | Brockbank (2013) | L | 1 |
| C$_5$H$_{10}$O$_2$ | $2.6\times10^{-2}$ | 5500 | Plyasunov et al. (2004) | L | |
| (methyl isobutyrate) | $3.3\times10^{-2}$ | 5700 | Bagno et al. (1991) | T | 473 |
| [547-63-7] | | 5700 | Della Gatta et al. (1981) | T | |
| BHIWKHZACMWKOJ-UHFFFAOYSA-N | | | | | |
| ethyl 2-methylpropanoate | $1.5\times10^{-2}$ | 6000 | Brockbank (2013) | L | 1 |
| C$_6$H$_{12}$O$_2$ | $1.7\times10^{-2}$ | 6200 | Plyasunov et al. (2004) | L | |
| (ethyl isobutyrate) | $2.0\times10^{-2}$ | | Hilal et al. (2008) | Q | |
| [97-62-1] | $1.8\times10^{-2}$ | | Yao et al. (2002) | Q | 229 |
| WDAXFOBOLVPGLV-UHFFFAOYSA-N | $2.4\times10^{-2}$ | | Yaws (1999) | ? | 21 |
| hexyl butanoate | $1.1\times10^{-2}$ | | Dupeux et al. (2022) | Q | 259 |
| C$_{10}$H$_{20}$O$_2$ | | | | | |
| (hexyl butyrate) | | | | | |
| [2639-63-6] | | | | | |
| XAPCMTMQBXLDBB-UHFFFAOYSA-N | | | | | |



Table A3.8: Esters (RCOOR) (... continued)

| Substance Formula (Trivial Name) [CAS Registry Number] InChIKey | $H_s^{cp}$ (at $T^\ominus$) $\left[\dfrac{\text{mol}}{\text{m}^3\,\text{Pa}}\right]$ | $\dfrac{\text{d}\ln H_s^{cp}}{\text{d}(1/T)}$ [K] | Reference | Type | Note |
|---|---|---|---|---|---|
| cyclohexyl butanoate $C_{10}H_{18}O_2$ (cyclohexyl butyrate) [1551-44-6] VZHUBBUZNIULNM-UHFFFAOYSA-N | | 6500 5600 | Kühne et al. (2005) Kühne et al. (2005) | Q ? | |
| 3-oxobutanoic acid, methyl ester $C_5H_8O_3$ (methylacetoacetate) [105-45-3] WRQNANDWMGAFTP-UHFFFAOYSA-N | 6.4 $3.6\times10^1$ $3.7\times10^1$ $1.2\times10^1$ $1.2\times10^1$ $2.5\times10^1$ $7.8\times10^1$ $1.7\times10^1$ 7.9 | | Hovorka et al. (2002) Duchowicz et al. (2020) HSDB (2015) Duchowicz et al. (2020) Raventos-Duran et al. (2010) Raventos-Duran et al. (2010) Raventos-Duran et al. (2010) Hilal et al. (2008) Modarresi et al. (2007) | M V V Q Q Q Q Q Q | 38 186 242, 243 244 245 67 |
| 3-oxobutanoic acid, ethyl ester $C_6H_{10}O_3$ (ethylacetoacetate) [141-97-9] XYIBRDXRRQCHLP-UHFFFAOYSA-N | 9.0 5.8 8.2 $1.7\times10^1$ $1.7\times10^1$ $2.4\times10^1$ 4.5 7.8 $2.0\times10^1$ $6.2\times10^1$ $1.7\times10^1$ $1.1\times10^1$ 5.6 9.1 | 7200 | Brockbank (2013) Hovorka et al. (2002) Duchowicz et al. (2020) Yaws (2003) Yaws (2003) Dupeux et al. (2022) Duchowicz et al. (2020) Raventos-Duran et al. (2010) Raventos-Duran et al. (2010) Raventos-Duran et al. (2010) Gharagheizi et al. (2010) Hilal et al. (2008) Modarresi et al. (2007) Yaws (1999) | L M V X X Q Q Q Q Q Q Q Q ? | 1 38 186 258 237, 154 259 271, 243 244 245 246 67 21, 154 |
| methyl pentanoate $C_4H_9COOCH_3$ (methyl valerate) [624-24-8] HNBDRPTVWVGKBR-UHFFFAOYSA-N | $3.0\times10^{-2}$ $3.1\times10^{-2}$ $9.7\times10^{-1}$ $3.9\times10^{-2}$ $2.4\times10^{-1}$ $2.9\times10^{-2}$ $3.1\times10^{-2}$ $2.0\times10^{-2}$ $2.5\times10^{-2}$ $2.2\times10^{-2}$ $3.6\times10^{-2}$ $3.1\times10^{-2}$ $3.8\times10^{-2}$ $2.6\times10^{-2}$ $2.5\times10^{-2}$ $3.1\times10^{-2}$ $3.1\times10^{-2}$ | 6100 5000 6200 | Plyasunov et al. (2004) Buttery et al. (1969) Djerki and Laub (1988) Della Gatta et al. (1981) Keshavarz et al. (2022) Duchowicz et al. (2020) Li et al. (2014) Raventos-Duran et al. (2010) Raventos-Duran et al. (2010) Raventos-Duran et al. (2010) Hilal et al. (2008) Modarresi et al. (2007) English and Carroll (2001) Katritzky et al. (1998) Suzuki et al. (1992) Nirmalakhandan and Speece (1988) Duchowicz et al. (2020) Abraham et al. (1990) | L M V T Q Q Q Q Q Q Q Q Q Q Q Q ? ? | 299 241 271, 243 244 245 67 230, 231 232 185, 21 |



Table A3.8: Esters (RCOOR) (...continued)

| Substance Formula (Trivial Name) [CAS Registry Number] InChIKey | $H_s^{cp}$ (at $T^\ominus$) $\left[\dfrac{\text{mol}}{\text{m}^3\,\text{Pa}}\right]$ | $\dfrac{\text{d}\ln H_s^{cp}}{\text{d}(1/T)}$ [K] | Reference | Type | Note |
|---|---|---|---|---|---|
| ethyl pentanoate | $2.3\times10^{-2}$ | 6800 | Plyasunov et al. (2004) | L | |
| $C_4H_9COOC_2H_5$ | $1.4\times10^{-1}$ | 4800 | Meynier et al. (2003) | M | 38 |
| (ethyl valerate) | $2.7\times10^{-2}$ | | Duchowicz et al. (2020) | V | 186 |
| [539-82-2] | $2.8\times10^{-2}$ | | Meylan and Howard (1991) | V | |
| ICMAFTSLXCXHRK-UHFFFAOYSA-N | $2.8\times10^{-2}$ | | Abraham (1984) | V | |
| | $2.9\times10^{-2}$ | | Hine and Mookerjee (1975) | V | |
| | $2.7\times10^{-2}$ | | Meynier et al. (2003) | C | |
| | $8.6\times10^{-2}$ | | Duchowicz et al. (2020) | Q | |
| | $2.5\times10^{-2}$ | | Raventos-Duran et al. (2010) | Q | 242, 243 |
| | $1.6\times10^{-2}$ | | Raventos-Duran et al. (2010) | Q | 244 |
| | $2.0\times10^{-2}$ | | Raventos-Duran et al. (2010) | Q | 245 |
| | $1.5\times10^{-2}$ | | Hilal et al. (2008) | Q | |
| | $2.6\times10^{-2}$ | | Modarresi et al. (2007) | Q | 67 |
| | $2.3\times10^{-2}$ | | English and Carroll (2001) | Q | 230, 231 |
| | $2.0\times10^{-2}$ | | Suzuki et al. (1992) | Q | 232 |
| | $1.8\times10^{-2}$ | | Meylan and Howard (1991) | Q | |
| | $2.2\times10^{-2}$ | | Nirmalakhandan and Speece (1988) | Q | |
| | $2.7\times10^{-2}$ | | Abraham et al. (1990) | ? | |
| methyl 2,2-dimethylpropanoate | $9.5\times10^{-3}$ | 5600 | Plyasunov et al. (2004) | L | |
| $C_6H_{12}O_2$ | $2.3\times10^{-2}$ | 6000 | Bagno et al. (1991) | T | 473 |
| (methyl pivalate) | | 6000 | Della Gatta et al. (1981) | T | |
| [598-98-1] | $3.9\times10^{-2}$ | | Keshavarz et al. (2022) | Q | |
| CNMFHDIDIMZHKY-UHFFFAOYSA-N | $4.1\times10^{-2}$ | | Duchowicz et al. (2020) | Q | |
| | $3.1\times10^{-2}$ | | Raventos-Duran et al. (2010) | Q | 242, 243 |
| | $1.6\times10^{-2}$ | | Raventos-Duran et al. (2010) | Q | 244 |
| | $2.5\times10^{-2}$ | | Raventos-Duran et al. (2010) | Q | 245 |
| | $1.7\times10^{-2}$ | | Hilal et al. (2008) | Q | |
| | $3.7\times10^{-2}$ | | Modarresi et al. (2007) | Q | 67 |
| | $1.8\times10^{-2}$ | | Yaffe et al. (2003) | Q | 248, 272 |
| | $2.3\times10^{-2}$ | | English and Carroll (2001) | Q | 230, 231 |
| | $1.8\times10^{-2}$ | | Nirmalakhandan et al. (1997) | Q | |
| | $2.3\times10^{-2}$ | | Duchowicz et al. (2020) | ? | 185, 21 |
| ethyl 2-methylbutanoate | $1.4\times10^{-2}$ | 6700 | Plyasunov et al. (2004) | L | |
| $C_7H_{14}O_2$ | $8.9\times10^{-3}$ | | Pollien et al. (2003) | M | |
| [7452-79-1] | $2.7\times10^{-2}$ | | Roberts and Pollien (1997) | M | |
| HCRBXQFHJMCTLF-UHFFFAOYSA-N | | | | | |
| ethyl 3-methylbutanoate | $1.4\times10^{-2}$ | 6500 | Brockbank (2013) | L | 1 |
| $C_7H_{14}O_2$ | $1.4\times10^{-2}$ | 6700 | Plyasunov et al. (2004) | L | |
| (ethyl isovalerate) | $1.1\times10^{-2}$ | 5900 | Wieland et al. (2015) | M | 513 |
| [108-64-5] | $1.4\times10^{-2}$ | | Duchowicz et al. (2020) | V | 186 |
| PPXUHEORWJQRHJ-UHFFFAOYSA-N | $2.2\times10^{-2}$ | | Dupeux et al. (2022) | Q | 259 |
| | $3.3\times10^{-2}$ | | Duchowicz et al. (2020) | Q | |
| | $2.5\times10^{-2}$ | | Raventos-Duran et al. (2010) | Q | 271, 243 |
| | $1.6\times10^{-2}$ | | Raventos-Duran et al. (2010) | Q | 244 |
| | $2.0\times10^{-2}$ | | Raventos-Duran et al. (2010) | Q | 245 |





Table A3.8: Esters (RCOOR) (. . . continued)

| Substance<br>Formula<br>(Trivial Name)<br>[CAS Registry Number]<br>InChIKey | $H_s^{cp}$<br>(at $T^\ominus$)<br>$\left[\dfrac{\text{mol}}{\text{m}^3\,\text{Pa}}\right]$ | $\dfrac{\text{d}\ln H_s^{cp}}{\text{d}(1/T)}$<br><br>[K] | Reference | Type | Note |
|---|---|---|---|---|---|
| | $1.6\times10^{-2}$ | | Hilal et al. (2008) | Q | |
| | $2.8\times10^{-2}$ | | Modarresi et al. (2007) | Q | 67 |
| | $1.5\times10^{-2}$ | | Yao et al. (2002) | Q | 229 |
| | $1.6\times10^{-2}$ | | Yaws (1999) | ? | 21 |
| ethyl 2,2-dimethylpropanoate<br>$C_7H_{14}O_2$<br>(ethyl pivalate)<br>[3938-95-2]<br>HHEIMYAXCOIQCJ-UHFFFAOYSA-N | $6.4\times10^{-3}$<br>$5.7\times10^{-3}$ | 6700<br>6100 | Brockbank (2013)<br>Plyasunov et al. (2004) | L<br>L | 1 |
| butyl pentanoate<br>$C_9H_{18}O_2$<br>(butyl valerate)<br>[591-68-4]<br>OKJADYKTJJGKDX-UHFFFAOYSA-N | $9.8\times10^{-3}$<br>$2.3\times10^{-2}$<br>$2.0\times10^{-2}$<br>$1.5\times10^{-2}$ | | Plyasunov et al. (2004)<br>Dupeux et al. (2022)<br>Yao et al. (2002)<br>Yaws (1999) | L<br>Q<br>Q<br>? | <br>259<br>229<br>21 |
| 3-methylbutyl 3-methylbutanoate<br>$C_{10}H_{20}O_2$<br>(isoamyl isovalerate)<br>[659-70-1]<br>XINCECQTMHSORG-UHFFFAOYSA-N | $4.0\times10^{-3}$ | | Yaws (1999) | ? | 21 |
| methyl hexanoate<br>$C_5H_{11}COOCH_3$<br>(methyl caproate)<br>[106-70-7]<br>NUKZAGXMHTUAFE-UHFFFAOYSA-N | $2.6\times10^{-2}$<br>$1.9\times10^{-2}$<br>$2.7\times10^{-2}$<br>$2.6$<br>$5.3\times10^{-2}$<br>$2.5\times10^{-1}$<br>$2.7\times10^{-2}$<br>$2.5\times10^{-2}$<br>$1.6\times10^{-2}$<br>$2.0\times10^{-2}$<br>$1.8\times10^{-2}$<br>$2.9\times10^{-2}$<br>$2.7\times10^{-2}$<br>$2.3\times10^{-2}$<br>$3.6\times10^{-2}$<br>$2.0\times10^{-2}$<br>$2.0\times10^{-2}$<br>$2.7\times10^{-2}$<br>$2.7\times10^{-2}$ | 6600<br><br><br>5600 | Plyasunov et al. (2004)<br>Aprea et al. (2007)<br>Buttery et al. (1969)<br>Djerki and Laub (1988)<br>Keshavarz et al. (2022)<br>Duchowicz et al. (2020)<br>Li et al. (2014)<br>Raventos-Duran et al. (2010)<br>Raventos-Duran et al. (2010)<br>Raventos-Duran et al. (2010)<br>Hilal et al. (2008)<br>Modarresi et al. (2007)<br>Yaffe et al. (2003)<br>English and Carroll (2001)<br>Katritzky et al. (1998)<br>Suzuki et al. (1992)<br>Nirmalakhandan and Speece (1988)<br>Duchowicz et al. (2020)<br>Abraham et al. (1990) | L<br>M<br>M<br>V<br>Q<br>Q<br>Q<br>Q<br>Q<br>Q<br>Q<br>Q<br>Q<br>Q<br>Q<br>Q<br>Q<br>?<br>? | <br><br><br><br><br>299<br>241<br>242, 243<br>244<br>245<br><br>67<br>248, 249<br>230, 274<br><br>232<br><br>185, 21<br> |
| ethyl hexanoate<br>$C_5H_{11}COOC_2H_5$<br>(ethyl caproate)<br>[123-66-0]<br>SHZIWNPUGXLXDT-UHFFFAOYSA-N | $1.9\times10^{-2}$<br>$1.4\times10^{-2}$<br>$1.6\times10^{-2}$<br>$9.4\times10^{-3}$<br>$6.9\times10^{-3}$<br>$1.8\times10^{-2}$<br>$3.0$ | 7200 | Plyasunov et al. (2004)<br>Aprea et al. (2007)<br>Landy et al. (1996)<br>Landy et al. (1995)<br>Philippe et al. (2003)<br>Abraham (1984)<br>Abney (2021) | L<br>M<br>M<br>M<br>V<br>V<br>Q | <br><br><br><br>14<br><br>399 |



Table A3.8: Esters (RCOOR) (...continued)

| Substance Formula (Trivial Name) [CAS Registry Number] InChIKey | $H_s^{cp}$ (at $T^\ominus$) $\left[\dfrac{\mathrm{mol}}{\mathrm{m}^3\,\mathrm{Pa}}\right]$ | $\dfrac{\mathrm{d}\ln H_s^{cp}}{\mathrm{d}(1/T)}$ [K] | Reference | Type | Note |
|---|---|---|---|---|---|
| | $1.4\times10^{-2}$ | | Savary et al. (2014) | Q | |
| | $1.1\times10^{-2}$ | | Hilal et al. (2008) | Q | |
| | $1.7\times10^{-2}$ | | English and Carroll (2001) | Q | 230, 260 |
| | $1.7\times10^{-2}$ | | Nirmalakhandan et al. (1997) | Q | |
| | $1.8\times10^{-2}$ | | Abraham et al. (1990) | ? | |
| 2-ethylbutanoic acid, 1,2-ethanediylbis(oxy-2,1-ethanediyl) ester $C_{18}H_{34}O_6$ [95-08-9] JEYLQCXBYFQJRO-UHFFFAOYSA-N | $9.9\times10^{5}$ | | HSDB (2015) | Q | 99 |
| methyl heptanoate $C_8H_{16}O_2$ [106-73-0] XNCNNDVCAUWAIT-UHFFFAOYSA-N | $1.8\times10^{-2}$ | | Plyasunov et al. (2004) | L | |
| ethyl heptanoate $C_6H_{13}COOC_2H_5$ [106-30-9] TVQGDYNRXLTQAP-UHFFFAOYSA-N | $2.0\times10^{-2}$ | | Meylan and Howard (1991) | V | |
| | $2.0\times10^{-2}$ | | Abraham (1984) | V | |
| | $2.0\times10^{-2}$ | | Hine and Mookerjee (1975) | V | |
| | $9.6\times10^{-2}$ | | Keshavarz et al. (2022) | Q | |
| | $9.1\times10^{-2}$ | | Duchowicz et al. (2020) | Q | 299 |
| | $1.2\times10^{-2}$ | | Raventos-Duran et al. (2010) | Q | 271, 243 |
| | $7.8\times10^{-3}$ | | Raventos-Duran et al. (2010) | Q | 244 |
| | $9.9\times10^{-3}$ | | Raventos-Duran et al. (2010) | Q | 245 |
| | $9.2\times10^{-3}$ | | Hilal et al. (2008) | Q | |
| | $1.9\times10^{-2}$ | | Modarresi et al. (2007) | Q | 67 |
| | $1.2\times10^{-2}$ | | Suzuki et al. (1992) | Q | 232 |
| | $1.0\times10^{-2}$ | | Meylan and Howard (1991) | Q | |
| | $2.1\times10^{-2}$ | | Nirmalakhandan and Speece (1988) | Q | |
| | $2.0\times10^{-2}$ | | Duchowicz et al. (2020) | ? | 185, 21 |
| | $2.0\times10^{-2}$ | | Abraham et al. (1990) | ? | |
| methyl octanoate $C_6H_{13}COOCH_3$ [111-11-5] JGHZJRVDZXSNKQ-UHFFFAOYSA-N | $1.2\times10^{-2}$ | | Plyasunov et al. (2004) | L | |
| | $9.9\times10^{-3}$ | | Aprea et al. (2007) | M | |
| | $1.3\times10^{-2}$ | | Buttery et al. (1969) | M | |
| | $9.6\times10^{-2}$ | | Keshavarz et al. (2022) | Q | |
| | $2.7\times10^{-1}$ | | Duchowicz et al. (2020) | Q | 299 |
| | $1.3\times10^{-2}$ | | Li et al. (2014) | Q | 241 |
| | $1.2\times10^{-2}$ | | Raventos-Duran et al. (2010) | Q | 242, 243 |
| | $1.2\times10^{-2}$ | | Raventos-Duran et al. (2010) | Q | 244 |
| | $9.9\times10^{-3}$ | | Raventos-Duran et al. (2010) | Q | 245 |
| | $1.2\times10^{-2}$ | | Hilal et al. (2008) | Q | |
| | $2.2\times10^{-2}$ | | Modarresi et al. (2007) | Q | 67 |
| | $4.7\times10^{-2}$ | | Nirmalakhandan et al. (1997) | Q | |
| | $1.2\times10^{-2}$ | | Suzuki et al. (1992) | Q | 232 |
| | $1.1\times10^{-2}$ | | Nirmalakhandan and Speece (1988) | Q | |
| | $1.3\times10^{-2}$ | | Duchowicz et al. (2020) | ? | 185, 21 |



Table A3.8: Esters (RCOOR) (...continued)

| Substance Formula (Trivial Name) [CAS Registry Number] InChIKey | $H_s^{cp}$ (at $T^{\ominus}$) $\left[\dfrac{\text{mol}}{\text{m}^3\,\text{Pa}}\right]$ | $\dfrac{\mathrm{d}\ln H_s^{cp}}{\mathrm{d}(1/T)}$ [K] | Reference | Type | Note |
|---|---|---|---|---|---|
| ethyl octanoate $C_7H_{15}COOC_2H_5$ [106-32-1] YYZUSRORWSJGET-UHFFFAOYSA-N | $1.7\times10^{-2}$ $1.1\times10^{-2}$ $1.2\times10^{-2}$ $7.8\times10^{-3}$ | | Plyasunov et al. (2004) Aprea et al. (2007) Abraham (1984) Savary et al. (2014) | L M V Q | |
| octadecanoic acid, 2-methylpropyl ester $C_{22}H_{44}O_2$ (isobutyl stearate) [646-13-9] ORFWYUFLWUWSFM-UHFFFAOYSA-N | $2.6\times10^{-4}$ | | HSDB (2015) | Q | 99 |
| octadecanoic acid, butyl ester $C_{22}H_{44}O_2$ [123-95-5] ULBTUVJTXULMLP-UHFFFAOYSA-N | $2.6\times10^{-4}$ | | HSDB (2015) | Q | 99 |
| methyl nonanoate $C_{10}H_{20}O_2$ [1731-84-6] IJXHLVMUNBOGRR-UHFFFAOYSA-N | $8.2\times10^{-3}$ $7.0\times10^{-3}$ | | Plyasunov et al. (2004) Abraham (1984) | L V | |
| ethyl nonanoate $C_8H_{17}COOC_2H_5$ [123-29-5] BYEVBITUADOIGY-UHFFFAOYSA-N | $1.3\times10^{-2}$ | | Abraham (1984) | V | |
| nonanedioic acid, bis(2-ethylhexyl) ester $C_{25}H_{48}O_4$ (di-2-ethylhexyl azelate) [103-24-2] ZDWGXBPVPXVXMQ-UHFFFAOYSA-N | $8.2\times10^{-2}$ | | HSDB (2015) | Q | 99 |
| methyl decanoate $C_{11}H_{22}O_2$ (methyl caprate) [110-42-9] YRHYCMZPEVDGFQ-UHFFFAOYSA-N | $1.1\times10^{-2}$ $4.8\times10^{-3}$ $1.4\times10^{-2}$ $3.2\times10^{-3}$ $2.8\times10^{-1}$ $5.8\times10^{-3}$ $7.7\times10^{-3}$ | | Aprea et al. (2007) Duchowicz et al. (2020) Krop et al. (1997) Abraham (1984) Duchowicz et al. (2020) HSDB (2015) Hilal et al. (2008) | M V V V Q Q Q | 186 99 |
| ethyl decanoate $C_9H_{19}COOC_2H_5$ [110-38-3] RGXWDWUGBIJHDO-UHFFFAOYSA-N | $1.2\times10^{-2}$ $1.4\times10^{-2}$ $1.7\times10^{-2}$ | | Plyasunov et al. (2004) Aprea et al. (2007) Abraham (1984) | L M V | |



Table A3.8: Esters (RCOOR) (...continued)

| Substance<br>Formula<br>(Trivial Name)<br>[CAS Registry Number]<br>InChIKey | $H_s^{cp}$<br>(at $T^\ominus$)<br><br>$\left[\dfrac{\text{mol}}{\text{m}^3\,\text{Pa}}\right]$ | $\dfrac{\mathrm{d}\ln H_s^{cp}}{\mathrm{d}(1/T)}$<br><br>[K] | Reference | Type | Note |
|---|---|---|---|---|---|
| decanedioic acid, diethyl ester<br>$C_{14}H_{26}O_4$<br>(diethyl sebacate)<br>[110-40-7]<br>ONKUXPIBXRRIDU-UHFFFAOYSA-N | 2.7 | | Bartelt-Hunt et al. (2008) | ? | 21 |
| methyl dodecanoate<br>$C_{13}H_{26}O_2$<br>(methyl laurate)<br>[111-82-0]<br>UQDUPQYQJKYHQI-UHFFFAOYSA-N | $8.3\times10^{-3}$<br>$7.5\times10^{-3}$<br>$3.3\times10^{-3}$<br>$4.8\times10^{-3}$ | | Krop et al. (1997)<br>Dupeux et al. (2022)<br>HSDB (2015)<br>Hilal et al. (2008) | V<br>Q<br>Q<br>Q | <br>259<br>99<br> |
| ethyl dodecanoate<br>$C_{14}H_{28}O_2$<br>(ethyl laurate)<br>[106-33-2]<br>MMXKVMNBHPAILY-UHFFFAOYSA-N | $7.7\times10^{-3}$<br>$3.1\times10^{-3}$ | | Krop et al. (1997)<br>Hilal et al. (2008) | V<br>Q | |
| propyl dodecanoate<br>$C_{15}H_{30}O_2$<br>(propyl laurate)<br>[3681-78-5]<br>FTBUKOLPOATXGV-UHFFFAOYSA-N | $7.7\times10^{-3}$<br>$2.1\times10^{-3}$ | | Krop et al. (1997)<br>Hilal et al. (2008) | V<br>Q | |
| 2-hydroxypropyl dodecanoate<br>$C_{15}H_{30}O_3$<br>[142-55-2]<br>BHIZVZJETFVJMJ-UHFFFAOYSA-N | $9.5\times10^{-2}$ | | Ebert et al. (2023) | ? | 318 |
| dodecanoic acid,<br>2-hydroxy-1-methylethyl ester<br>$C_{15}H_{30}O_3$<br>[107328-11-0]<br>SVWZGNLBKFWCMV-UHFFFAOYSA-N | $2.6\times10^{1}$ | | Ebert et al. (2023) | ? | 318 |
| butyl dodecanoate<br>$C_{16}H_{32}O_2$<br>(butyl laurate)<br>[106-18-3]<br>NDKYEUQMPZIGFN-UHFFFAOYSA-N | $7.1\times10^{-3}$<br>$1.5\times10^{-3}$ | | Krop et al. (1997)<br>Hilal et al. (2008) | V<br>Q | |
| 2-ethylhexyl dodecanoate<br>$C_{20}H_{40}O_2$<br>(2-ethylhexyl laurate)<br>[20292-08-4]<br>LWLRMRFJCCMNML-UHFFFAOYSA-N | $3.0\times10^{-3}$<br>$8.6\times10^{-4}$ | | Krop et al. (1997)<br>Hilal et al. (2008) | V<br>Q | |





Table A3.8: Esters (RCOOR) (. . . continued)

| Substance<br>Formula<br>(Trivial Name)<br>[CAS Registry Number]<br>InChIKey | $H_s^{cp}$<br>(at $T^{\ominus}$)<br>$\left[\dfrac{\text{mol}}{\text{m}^3\,\text{Pa}}\right]$ | $\dfrac{\text{d}\ln H_s^{cp}}{\text{d}(1/T)}$<br><br>[K] | Reference | Type | Note |
|---|---|---|---|---|---|
| methyl tetradecanoate<br>$C_{15}H_{30}O_2$<br>(methyl myristate)<br>[124-10-7]<br>ZAZKJZBWRNNLDS-UHFFFAOYSA-N | $5.0\times10^{-3}$<br>$1.9\times10^{-3}$<br>$3.1\times10^{-3}$ | | Krop et al. (1997)<br>HSDB (2015)<br>Hilal et al. (2008) | V<br>Q<br>Q | <br>99<br> |
| methyl hexadecanoate<br>$C_{17}H_{34}O_2$<br>(methyl palmitate)<br>[112-39-0]<br>FLIACVVOZYBSBS-UHFFFAOYSA-N | $2.9\times10^{-3}$<br>$1.1\times10^{-3}$<br>$1.8\times10^{-3}$ | | Krop et al. (1997)<br>HSDB (2015)<br>Hilal et al. (2008) | V<br>Q<br>Q | <br>99<br> |
| isopropyl palmitate<br>$C_{19}H_{38}O_2$<br>[142-91-6]<br>XUGNVMKQXJXZCD-UHFFFAOYSA-N | $2.1\times10^{-4}$ | | HSDB (2015) | Q | 447 |
| ascorbic palmitate<br>$C_{22}H_{38}O_7$<br>[137-66-6]<br>QAQJMLQRFWZOBN-LAUBAEHRSA-N | $7.0\times10^{1}$ | | HSDB (2015) | Q | 99 |
| methyl octadecanoate<br>$C_{19}H_{38}O_2$<br>(methyl stearate)<br>[112-61-8]<br>HPEUJPJOZXNMSJ-UHFFFAOYSA-N | $1.7\times10^{-3}$<br>$6.2\times10^{-4}$<br>$1.1\times10^{-3}$ | | Krop et al. (1997)<br>HSDB (2015)<br>Hilal et al. (2008) | V<br>Q<br>Q | <br>99<br> |
| methyl eicosanoate<br>$C_{21}H_{42}O_2$<br>(methyl arachidate)<br>[1120-28-1]<br>QGBRLVONZXHAKJ-UHFFFAOYSA-N | $1.0\times10^{-3}$ | | Krop et al. (1997) | V | |
| methyl docosanoate<br>$C_{23}H_{46}O_2$<br>(methyl behenate)<br>[929-77-1]<br>QSQLTHHMFHEFIY-UHFFFAOYSA-N | $5.9\times10^{-4}$ | | Krop et al. (1997) | V | |
| cyclopropanecarboxylic acid,<br>methyl ester<br>$C_5H_8O_2$<br>[2868-37-3]<br>PKAHQJNJPDVTDP-UHFFFAOYSA-N | $4.1\times10^{-1}$<br><br>$1.1\times10^{-1}$<br>$1.4\times10^{-1}$ | 6100 | Bagno et al. (1991)<br><br>Hilal et al. (2008)<br>English and Carroll (2001) | T<br><br>Q<br>Q | 473<br><br><br>230, 231 |
| cyclohexanecarboxylic acid,<br>methyl ester<br>$C_6H_{11}COOCH_3$<br>[4630-82-4]<br>ZQWPRMPSCMSAJU-UHFFFAOYSA-N | $1.1\times10^{-1}$<br><br>$1.1\times10^{-1}$ | 7200 | Bagno et al. (1991)<br><br>English and Carroll (2001) | T<br><br>Q | 473<br><br>230, 231 |



Table A3.8: Esters (RCOOR) (...continued)

| Substance<br>Formula<br>(Trivial Name)<br>[CAS Registry Number]<br>InChIKey | $H_s^{cp}$<br>(at $T^{\ominus}$)<br>$\left[\dfrac{\mathrm{mol}}{\mathrm{m}^3\,\mathrm{Pa}}\right]$ | $\dfrac{\mathrm{d}\ln H_s^{cp}}{\mathrm{d}(1/T)}$<br><br>[K] | Reference | Type | Note |
|---|---|---|---|---|---|
| ($Z,Z,Z$)-9,12,15-octadecatrienoic acid, methyl ester<br>$C_{19}H_{32}O_2$<br>(methyl linolenate)<br>[301-00-8]<br>DVWSXZIHSUZZKJ-YSTUJMKBSA-N | $2.8\times10^{-1}$<br><br>$7.2\times10^{-3}$ | | Krop et al. (1997)<br><br>Hilal et al. (2008) | V<br><br>Q | |
| ($Z,Z$)-9,12-octadecadienoic acid, methyl ester<br>$C_{19}H_{34}O_2$<br>(methyl linolate)<br>[112-63-0]<br>WTTJVINHCBCLGX-NQLNTKRDSA-N | $6.2\times10^{-2}$<br><br>$4.8\times10^{-3}$ | | Krop et al. (1997)<br><br>Hilal et al. (2008) | V<br><br>Q | |
| ($Z$)-9-octadecenoic acid, methyl ester<br>$C_{19}H_{36}O_2$<br>(methyl oleate)<br>[112-62-9]<br>QYDYPVFESGNLHU-KHPPLWFESA-N | $1.3\times10^{-2}$<br><br>$7.0\times10^{-4}$<br>$2.5\times10^{-3}$ | | Krop et al. (1997)<br><br>HSDB (2015)<br>Hilal et al. (2008) | V<br><br>Q<br>Q | <br><br>99 |
| ($Z$)-13-docosenoic acid, methyl ester<br>$C_{23}H_{44}O_2$<br>(methyl erucate)<br>[1120-34-9]<br>ZYNDJIBBPLNPOW-KHPPLWFESA-N | $5.3\times10^{-3}$<br><br>$8.2\times10^{-4}$ | | Krop et al. (1997)<br><br>Hilal et al. (2008) | V<br><br>Q | |
| *trans*-quercus lactone<br>$C_9H_{16}O_2$<br>(*trans*-whisky lactone)<br>[105119-22-0]<br>WNVCMFHPRIBNCW-SFYZADRCSA-N | $3.5\times10^{1}$ | | Abney (2021) | Q | 399 |
| $\gamma$-decalactone<br>$C_{10}H_{18}O_2$<br>[706-14-9]<br>IFYYFLINQYPWGJ-UHFFFAOYSA-N | $5.3\times10^{1}$ | | Abney (2021) | Q | 399 |
| $\gamma$-dodecalactone<br>$C_{12}H_{22}O_2$<br>[2305-05-7]<br>WGPCZPLRVAWXPW-UHFFFAOYSA-N | $3.7\times10^{1}$ | | Abney (2021) | Q | 399 |
| oxacyclohexadecan-2-one<br>$C_{15}H_{28}O_2$<br>(pentadecalactone)<br>[106-02-5]<br>FKUPPRZPSYCDRS-UHFFFAOYSA-N | $4.0\times10^{-3}$<br>$7.6\times10^{-2}$ | | Amoore and Buttery (1978)<br>Amoore and Buttery (1978) | M<br>V | |



Table A3.8: Esters (RCOOR) (...continued)

| Substance<br>Formula<br>(Trivial Name)<br>[CAS Registry Number]<br>InChIKey | $H_s^{cp}$<br>(at $T^{\ominus}$)<br>$\left[\dfrac{\mathrm{mol}}{\mathrm{m^3\,Pa}}\right]$ | $\dfrac{\mathrm{d}\ln H_s^{cp}}{\mathrm{d}(1/T)}$<br><br>[K] | Reference | Type | Note |
|---|---|---|---|---|---|
| 2-*tert*-butylcyclohexyl acetate<br>$C_{12}H_{22}O_2$<br>[88-41-5]<br>FINOAUDUYKVGDS-UHFFFAOYSA-N | $9.9\times10^{-3}$<br>$3.8\times10^{-2}$<br>$5.3\times10^{-1}$<br>$7.0\times10^{-3}$ | | Zhang et al. (2010)<br>Zhang et al. (2010)<br>Zhang et al. (2010)<br>Zhang et al. (2010) | Q<br>Q<br>Q<br>Q | 287, 288<br>287, 289<br>287, 290<br>287, 291 |
| hedione<br>$C_{13}H_{22}O_3$<br>[24851-98-7]<br>KVWWIYGFBYDJQC-UHFFFAOYSA-N | $7.0\times10^{1}$ | | Dupeux et al. (2022) | Q | 259 |
| ethylene brassylate<br>$C_{15}H_{26}O_4$<br>[105-95-3]<br>XRHCAGNSDHCHFJ-UHFFFAOYSA-N | $5.3\times10^{1}$ | | Dupeux et al. (2022) | Q | 259 |
| propyl<br>3-oxo-2-pentylcyclopentaneacetate<br>$C_{15}H_{26}O_3$<br>[158474-72-7]<br>IPDFPNNPBMREIF-UHFFFAOYSA-N | $1.4\times10^{1}$ | | Ebert et al. (2023) | ? | 318 |
| 2-ethyl-3-oxo-butanoic acid, ethyl ester<br>$C_8H_{14}O_3$<br>[607-97-6]<br>OKANYBNORCUPKZ-UHFFFAOYSA-N | $3.4$ | | Hilal et al. (2008) | Q | |
| 2-hydroxypropanoic acid, butyl ester<br>$C_7H_{14}O_3$<br>[138-22-7]<br>MRABAEUHTLLEML-UHFFFAOYSA-N | $4.9$<br><br>$2.7$<br>$2.9\times10^{1}$<br>$2.6$<br>$1.6$<br>$1.2\times10^{1}$<br>$1.2\times10^{-1}$<br>$6.4\times10^{1}$<br>$4.3\times10^{1}$ | | HSDB (2015)<br><br>Wang et al. (2017)<br>Wang et al. (2017)<br>Wang et al. (2017)<br>Raventos-Duran et al. (2010)<br>Raventos-Duran et al. (2010)<br>Raventos-Duran et al. (2010)<br>Hilal et al. (2008)<br>Modarresi et al. (2007) | V<br><br>Q<br>Q<br>Q<br>Q<br>Q<br>Q<br>Q<br>Q | <br><br>80, 238<br>80, 239<br>80, 240<br>271, 243<br>244<br>245<br><br>67 |
| methyl propenoate<br>$C_4H_6O_2$<br>(methyl acrylate)<br>[96-33-3]<br>BAPJBEWLBFYGME-UHFFFAOYSA-N | $5.0\times10^{-2}$<br>$4.9\times10^{-2}$<br>$5.2\times10^{-2}$<br>$5.2\times10^{-2}$<br>$5.8\times10^{-2}$<br>$6.3\times10^{-1}$<br>$7.2\times10^{-2}$<br>$6.2\times10^{-2}$<br>$6.2\times10^{-2}$<br>$9.9\times10^{-2}$<br>$6.0\times10^{-2}$<br>$5.4\times10^{-2}$ | | Duchowicz et al. (2020)<br>HSDB (2015)<br>Mackay et al. (2006c)<br>Mackay et al. (1995)<br>Yaws (2003)<br>Duchowicz et al. (2020)<br>Gharagheizi et al. (2012)<br>Raventos-Duran et al. (2010)<br>Raventos-Duran et al. (2010)<br>Raventos-Duran et al. (2010)<br>Gharagheizi et al. (2010)<br>Hilal et al. (2008) | V<br>V<br>V<br>V<br>X<br>Q<br>Q<br>Q<br>Q<br>Q<br>Q<br>Q | 186<br><br><br><br>237, 72<br><br><br>242, 243<br>244<br>245<br>246<br> |



Table A3.8: Esters (RCOOR) (...continued)

| Substance Formula (Trivial Name) [CAS Registry Number] InChIKey | $H_s^{cp}$ (at $T^\ominus$) $\left[\dfrac{\text{mol}}{\text{m}^3\,\text{Pa}}\right]$ | $\dfrac{\text{d}\ln H_s^{cp}}{\text{d}(1/T)}$ [K] | Reference | Type | Note |
|---|---|---|---|---|---|
| | $7.4\times10^{-2}$ | | Modarresi et al. (2007) | Q | 67 |
| | $5.2\times10^{-2}$ | | Yaws (1999) | ? | 21, 72 |
| ethyl propenoate | $3.9\times10^{-2}$ | 4400 | Brockbank (2013) | L | 1 |
| $C_5H_8O_2$ | $2.9\times10^{-2}$ | | Duchowicz et al. (2020) | V | 186 |
| (ethyl acrylate) | $2.9\times10^{-2}$ | | HSDB (2015) | V | |
| [140-88-5] | $2.9\times10^{-2}$ | | Mackay et al. (2006c) | V | |
| JIGUQPWFLRLWPJ-UHFFFAOYSA-N | $2.9\times10^{-2}$ | | Mackay et al. (1995) | V | |
| | $3.9\times10^{-2}$ | | Yaws (2003) | X | 258 |
| | $4.7\times10^{-2}$ | | Dupeux et al. (2022) | Q | 259 |
| | $2.5\times10^{-1}$ | | Duchowicz et al. (2020) | Q | |
| | $8.8\times10^{-2}$ | | Gharagheizi et al. (2012) | Q | |
| | $4.9\times10^{-2}$ | | Raventos-Duran et al. (2010) | Q | 242, 243 |
| | $3.9\times10^{-2}$ | | Raventos-Duran et al. (2010) | Q | 244 |
| | $7.8\times10^{-2}$ | | Raventos-Duran et al. (2010) | Q | 245 |
| | $3.5\times10^{-2}$ | | Hilal et al. (2008) | Q | |
| | $5.3\times10^{-2}$ | | Modarresi et al. (2007) | Q | 67 |
| | $8.8\times10^{-2}$ | | Yao et al. (2002) | Q | 229 |
| | $4.0\times10^{-2}$ | | Yaws (1999) | ? | 21 |
| 2-propenoic acid, butyl ester | $1.5\times10^{-2}$ | | Duchowicz et al. (2020) | V | 186 |
| $C_7H_{12}O_2$ | $2.1\times10^{-2}$ | | HSDB (2015) | V | |
| (butyl acrylate) | $2.9\times10^{-1}$ | | Duchowicz et al. (2020) | Q | |
| [141-32-2] | $2.6\times10^{-2}$ | | Gharagheizi et al. (2012) | Q | |
| CQEYYJKEWSMYFG-UHFFFAOYSA-N | $2.5\times10^{-2}$ | | Raventos-Duran et al. (2010) | Q | 242, 243 |
| | $2.0\times10^{-2}$ | | Raventos-Duran et al. (2010) | Q | 244 |
| | $4.9\times10^{-2}$ | | Raventos-Duran et al. (2010) | Q | 245 |
| | $2.0\times10^{-2}$ | | Hilal et al. (2008) | Q | |
| | $4.4\times10^{-2}$ | | Modarresi et al. (2007) | Q | 67 |
| | $5.2\times10^{-2}$ | | Yao et al. (2002) | Q | 229, 267 |
| | $1.7\times10^{-2}$ | | Yaws (1999) | ? | 21, 12 |
| 2-propenoic acid, 2-methylpropyl ester | $1.3\times10^{-2}$ | | Duchowicz et al. (2020) | V | 186 |
| $C_7H_{12}O_2$ | $1.6\times10^{-2}$ | | HSDB (2015) | V | |
| (isobutyl acrylate) | $1.1\times10^{-1}$ | | Duchowicz et al. (2020) | Q | |
| [106-63-8] | $1.4\times10^{-2}$ | | Gharagheizi et al. (2012) | Q | |
| CFVWNXQPGQOHRJ-UHFFFAOYSA-N | $2.5\times10^{-2}$ | | Raventos-Duran et al. (2010) | Q | 242, 243 |
| | $2.5\times10^{-2}$ | | Raventos-Duran et al. (2010) | Q | 244 |
| | $4.9\times10^{-2}$ | | Raventos-Duran et al. (2010) | Q | 245 |
| | $2.4\times10^{-2}$ | | Hilal et al. (2008) | Q | |
| | $3.4\times10^{-2}$ | | Modarresi et al. (2007) | Q | 67 |
| | $3.9\times10^{-2}$ | | Yao et al. (2002) | Q | 229 |
| | $1.5\times10^{-2}$ | | Yaws (1999) | ? | 21, 72 |



Table A3.8: Esters (RCOOR) (...continued)

| Substance<br>Formula<br>(Trivial Name)<br>[CAS Registry Number]<br>InChIKey | $H_s^{cp}$<br>(at $T^\ominus$)<br>$\left[\dfrac{\text{mol}}{\text{m}^3\,\text{Pa}}\right]$ | $\dfrac{\mathrm{d}\ln H_s^{cp}}{\mathrm{d}(1/T)}$<br><br>[K] | Reference | Type | Note |
|---|---|---|---|---|---|
| 2-propenoic acid, 2-ethylhexyl ester | $2.3\times10^{-2}$ | | Duchowicz et al. (2020) | V | 186 |
| $C_{11}H_{20}O_2$ | $2.3\times10^{-2}$ | | HSDB (2015) | V | |
| (2-ethylhexyl acrylate) | $2.7\times10^{-2}$ | | Yaws (2003) | X | 237, 72 |
| [103-11-7] | $1.3\times10^{-1}$ | | Duchowicz et al. (2020) | Q | |
| GOXQRTZXKQZDDN-UHFFFAOYSA-N | $6.2\times10^{-3}$ | | Gharagheizi et al. (2012) | Q | |
| | $7.8\times10^{-3}$ | | Raventos-Duran et al. (2010) | Q | 271, 243 |
| | $1.2\times10^{-2}$ | | Raventos-Duran et al. (2010) | Q | 244 |
| | $1.6\times10^{-2}$ | | Raventos-Duran et al. (2010) | Q | 245 |
| | $2.8\times10^{-2}$ | | Gharagheizi et al. (2010) | Q | 246 |
| | $1.2\times10^{-2}$ | | Hilal et al. (2008) | Q | |
| | $2.2\times10^{-2}$ | | Modarresi et al. (2007) | Q | 67 |
| | $2.0\times10^{-2}$ | | Yao et al. (2002) | Q | 229 |
| | $2.3\times10^{-2}$ | | Yaws (1999) | ? | 21, 72 |
| 2-propenoic acid, 2-hydroxyethyl ester<br>$C_5H_8O_3$<br>(2-hydroxyethyl acrylate)<br>[818-61-1]<br>OMIGHNLMNHATMP-UHFFFAOYSA-N | $1.2\times10^{3}$ | | HSDB (2015) | V | |
| 2-methyl-2-propenoic acid, ethyl ester | $1.7\times10^{-2}$ | | Duchowicz et al. (2020) | V | 186 |
| $C_6H_{10}O_2$ | $1.7\times10^{-2}$ | | HSDB (2015) | V | |
| [97-63-2] | $1.6\times10^{-2}$ | | Hilal et al. (2008) | C | |
| SUPCQIBBMFXVTL-UHFFFAOYSA-N | $8.9\times10^{-2}$ | | Duchowicz et al. (2020) | Q | |
| | $3.1\times10^{-2}$ | | Raventos-Duran et al. (2010) | Q | 242, 243 |
| | $3.1\times10^{-2}$ | | Raventos-Duran et al. (2010) | Q | 244 |
| | $4.9\times10^{-2}$ | | Raventos-Duran et al. (2010) | Q | 245 |
| | $2.9\times10^{-2}$ | | Hilal et al. (2008) | Q | |
| | $3.5\times10^{-2}$ | | Modarresi et al. (2007) | Q | 67 |
| 2-methyl-2-propenoic acid, 2-propenyl ester<br>$C_7H_{10}O_2$<br>(allyl methacrylate)<br>[96-05-9]<br>FBCQUCJYYPMKRO-UHFFFAOYSA-N | $2.4\times10^{-2}$ | | HSDB (2015) | Q | 99 |
| 2-methyl-2-propenoic acid, oxiranylmethyl ester<br>$C_7H_{10}O_3$<br>(glycidyl methacrylate)<br>[106-91-2]<br>VOZRXNHHFUQHIL-UHFFFAOYSA-N | $3.2\times10^{1}$ | | HSDB (2015) | Q | 99 |





Table A3.8: Esters (RCOOR) (...continued)

| Substance<br>Formula<br>(Trivial Name)<br>[CAS Registry Number]<br>InChIKey | $H_s^{cp}$<br>(at $T^{\ominus}$)<br>$\left[\dfrac{\text{mol}}{\text{m}^3\,\text{Pa}}\right]$ | $\dfrac{\mathrm{d}\ln H_s^{cp}}{\mathrm{d}(1/T)}$<br><br>[K] | Reference | Type | Note |
|---|---|---|---|---|---|
| allyl acetoacetate<br>$C_7H_{10}O_3$<br>[1118-84-9]<br>AXLMPTNTPOWPLT-UHFFFAOYSA-N | $1.2\times10^1$ | | Ebert et al. (2023) | ? | 318 |
| 2-methyl-2-propenoic acid, propyl ester<br>$C_7H_{12}O_2$<br>(propyl methacrylate)<br>[2210-28-8]<br>NHARPDSAXCBDDR-UHFFFAOYSA-N | $1.8\times10^{-2}$ | | HSDB (2015) | Q | 99 |
| 2-methyl-2-propenoic acid, butyl ester<br>$C_8H_{14}O_2$<br>(butyl methacrylate)<br>[97-88-1]<br>SOGAXMICEFXMKE-UHFFFAOYSA-N | $2.0\times10^{-2}$<br>$2.0\times10^{-2}$<br>$9.9\times10^{-2}$<br>$2.0\times10^{-2}$<br>$1.6\times10^{-2}$<br>$3.1\times10^{-2}$<br>$1.8\times10^{-2}$<br>$4.0\times10^{-2}$ | | Duchowicz et al. (2020)<br>HSDB (2015)<br>Duchowicz et al. (2020)<br>Raventos-Duran et al. (2010)<br>Raventos-Duran et al. (2010)<br>Raventos-Duran et al. (2010)<br>Hilal et al. (2008)<br>Modarresi et al. (2007) | V<br>V<br>Q<br>Q<br>Q<br>Q<br>Q<br>Q | 186<br><br><br>242, 243<br>244<br>245<br><br>67 |
| 2-methyl-2-propenoic acid, 2-methylpropyl ester<br>$C_8H_{14}O_2$<br>[97-86-9]<br>RUMACXVDVNRZJZ-UHFFFAOYSA-N | $1.9\times10^{-2}$<br>$1.9\times10^{-2}$<br>$3.9\times10^{-2}$<br>$2.0\times10^{-2}$<br>$2.0\times10^{-2}$<br>$3.1\times10^{-2}$<br>$2.1\times10^{-2}$<br>$2.7\times10^{-2}$ | | Duchowicz et al. (2020)<br>HSDB (2015)<br>Duchowicz et al. (2020)<br>Raventos-Duran et al. (2010)<br>Raventos-Duran et al. (2010)<br>Raventos-Duran et al. (2010)<br>Hilal et al. (2008)<br>Modarresi et al. (2007) | V<br>V<br>Q<br>Q<br>Q<br>Q<br>Q<br>Q | 186<br><br><br>271, 243<br>244<br>245<br><br>67 |
| 2-methyl-2-propenoic acid, 1,2-ethanediylbis(oxy-2,1-ethanediyl) ester<br>$C_{14}H_{22}O_6$<br>[109-16-0]<br>HWSSEYVMGDIFMH-UHFFFAOYSA-N | $5.8\times10^6$ | | HSDB (2015) | Q | 99 |
| methyl methacrylate<br>$C_5H_8O_2$<br>[80-62-6]<br>VVQNEPGJFQJSBK-UHFFFAOYSA-N | $2.9\times10^{-2}$<br>$4.3\times10^{-2}$<br>$3.1\times10^{-2}$<br>$3.1\times10^{-2}$<br>$3.1\times10^{-2}$<br>$3.1\times10^{-2}$<br>$3.0\times10^{-2}$<br>$3.1\times10^{-2}$<br>$2.4\times10^{-1}$<br>$4.9\times10^{-2}$<br>$4.9\times10^{-2}$ | 5300<br>7700<br><br><br>5300 | Brockbank (2013)<br>Hiatt (2013)<br>Duchowicz et al. (2020)<br>HSDB (2015)<br>Dohnal et al. (2010)<br>Mackay et al. (2006c)<br>Lide and Frederikse (1995)<br>Mackay et al. (1995)<br>Duchowicz et al. (2020)<br>Raventos-Duran et al. (2010)<br>Raventos-Duran et al. (2010) | L<br>M<br>V<br>V<br>V<br>V<br>V<br>V<br>Q<br>Q<br>Q | 1<br><br>186<br><br>1<br><br><br><br><br>271, 243<br>244 |



Table A3.8: Esters (RCOOR) (...continued)

| Substance Formula (Trivial Name) [CAS Registry Number] InChIKey | $H_s^{cp}$ (at $T^{\ominus}$) $\left[\dfrac{\text{mol}}{\text{m}^3\,\text{Pa}}\right]$ | $\dfrac{\text{d}\ln H_s^{cp}}{\text{d}(1/T)}$ [K] | Reference | Type | Note |
|---|---|---|---|---|---|
| | $6.2\times10^{-2}$ | | Raventos-Duran et al. (2010) | Q | 245 |
| | $4.4\times10^{-2}$ | | Hilal et al. (2008) | Q | |
| | $3.5\times10^{-2}$ | | Modarresi et al. (2007) | Q | 67 |
| | | | Burkholder et al. (2019) | W | 514 |
| | | | Burkholder et al. (2015) | W | 515 |
| (E)-3-hexenyl ethanoate $C_8H_{14}O_2$ [3681-82-1] NPFVOOAXDOBMCE-SNAWJCMRSA-N | $3.3\times10^{-2}$ | | Karl et al. (2003) | M | |
| (Z)-3-hexenyl ethanoate $C_8H_{14}O_2$ [3681-71-8] NPFVOOAXDOBMCE-PLNGDYQASA-N | $3.1\times10^{-2}$ | | Karl et al. (2003) | M | |
| ethenyl ethanoate $CH_3COOCHCH_2$ (vinyl acetate) [108-05-4] XTXRWKRVRITETP-UHFFFAOYSA-N | $2.0\times10^{-2}$ | 4500 | Burkholder et al. (2019) | L | |
| | $2.0\times10^{-2}$ | 4500 | Burkholder et al. (2015) | L | |
| | $2.0\times10^{-2}$ | 4600 | Brockbank (2013) | L | 1 |
| | $2.0\times10^{-2}$ | 4400 | Böhme et al. (2008) | M | |
| | $1.9\times10^{-2}$ | | HSDB (2015) | V | |
| | $2.0\times10^{-2}$ | 4600 | Dohnal et al. (2010) | V | 1 |
| | $1.6\times10^{-2}$ | | Mackay et al. (2006c) | V | |
| | $2.0\times10^{-2}$ | | Lide and Frederikse (1995) | V | |
| | $1.6\times10^{-2}$ | | Mackay et al. (1995) | V | |
| | $2.0\times10^{-2}$ | | Yaws (2003) | X | 258 |
| | $2.0\times10^{-2}$ | | Yaws (2003) | X | 237 |
| | $1.7\times10^{-2}$ | | Goldstein (1982) | X | 446 |
| | $1.7\times10^{-2}$ | 2600 | Goldstein (1982) | X | 298 |
| | $1.2\times10^{-2}$ | | Dupeux et al. (2022) | Q | 259 |
| | $5.8\times10^{-2}$ | | Gharagheizi et al. (2012) | Q | |
| | $6.2\times10^{-2}$ | | Raventos-Duran et al. (2010) | Q | 242, 243 |
| | $7.8\times10^{-2}$ | | Raventos-Duran et al. (2010) | Q | 244 |
| | $7.8\times10^{-3}$ | | Raventos-Duran et al. (2010) | Q | 245 |
| | $2.2\times10^{-2}$ | | Gharagheizi et al. (2010) | Q | 246 |
| | $6.9\times10^{-2}$ | | Hilal et al. (2008) | Q | |
| | $6.6\times10^{-2}$ | | Modarresi et al. (2007) | Q | 67 |
| | $2.0\times10^{-2}$ | | Yaws (1999) | ? | 21 |
| (7E,9Z)-dodecadienyl acetate $C_{14}H_{24}O_2$ [54364-62-4] LLRZUAWETKPZJO-SCFJQAPRSA-N | $1.2\times10^{-2}$ | | Ebert et al. (2023) | ? | 318 |
| (9Z,12E)-9,12-tetradecadienyl acetate $C_{16}H_{28}O_2$ [30507-70-1] ZZGJZGSVLNSDPG-FDTUMDBZSA-N | $1.7\times10^{-1}$ | | Ebert et al. (2023) | ? | 318 |





Table A3.8: Esters (RCOOR) (...continued)

| Substance Formula (Trivial Name) [CAS Registry Number] InChIKey | $H_s^{cp}$ (at $T^{\ominus}$) $\left[\dfrac{\text{mol}}{\text{m}^3\,\text{Pa}}\right]$ | $\dfrac{\text{d}\ln H_s^{cp}}{\text{d}(1/T)}$ [K] | Reference | Type | Note |
|---|---|---|---|---|---|
| isoambrettolide $C_{16}H_{28}O_2$ [28645-51-4] QILMAYXCYBTEDM-IWQZZHSRSA-N | 2.5 | | Dupeux et al. (2022) | Q | 259 |
| vetyveryl acetate $C_{17}H_{26}O_2$ [117-98-6] UAVFEMBKDRODDE-UHFFFAOYSA-N | 2.2 | | Dupeux et al. (2022) | Q | 259 |
| hydroprene $C_{17}H_{30}O_2$ [41096-46-2] FYQGBXGJFWXIPP-UEVLXMDPSA-N | $5.1\times10^{-2}$ | | Ebert et al. (2023) | ? | 316 |
| empenthrin $C_{18}H_{26}O_2$ [54406-48-3] YUGWDVYLFSETPE-UHFFFAOYSA-N | $2.9\times10^{-2}$ | | Ebert et al. (2023) | ? | 316 |
| 3-(4-methoxyphenyl)-2-propenoic acid, 2-ethylhexyl ester $C_{18}H_{26}O_3$ (octinoxate) [5466-77-3] YBGZDTIWKVFICR-JLHYYAGUSA-N | 1.2 | | HSDB (2015) | Q | 447 |
| kinoprene $C_{18}H_{28}O_2$ [42588-37-4] FZRBKIRIBLNOAM-WHVZTFIZSA-N | $1.8\times10^{-1}$ | | Ebert et al. (2023) | ? | 316 |
| allethrin $C_{19}H_{26}O_3$ [584-79-2] ZCVAOQKBXKSDMS-UHFFFAOYSA-N | $6.0\times10^{1}$ | | Ebert et al. (2023) | ? | 318 |
| S-methoprene $C_{19}H_{34}O_3$ [65733-16-6] NFGXHKASABOEEW-GYMWBFJFSA-N | 1.5 | | Ebert et al. (2023) | ? | 318 |
| methyl benzoate $C_6H_5COOCH_3$ [93-58-3] QPJVMBTYPHYUOC-UHFFFAOYSA-N | $2.9\times10^{-1}$ | 6400 | Brockbank (2013) | L | 1 |
| | $3.0\times10^{-1}$ | | Duchowicz et al. (2020) | V | 186 |
| | $3.0\times10^{-1}$ | | HSDB (2015) | V | |
| | $3.0\times10^{-1}$ | | Mackay et al. (2006c) | V | |
| | $3.0\times10^{-1}$ | | Mackay et al. (1995) | V | |
| | $2.8\times10^{-1}$ | | Meylan and Howard (1991) | V | |
| | $5.6\times10^{-1}$ | | Hine and Mookerjee (1975) | V | |
| | $3.1\times10^{-1}$ | | Abraham et al. (1994a) | R | |
| | $5.8\times10^{-1}$ | 6300 | Bagno et al. (1991) | T | 473 |



Table A3.8: Esters (RCOOR) (...continued)

| Substance Formula (Trivial Name) [CAS Registry Number] InChIKey | $H_s^{cp}$ (at $T^\ominus$) $\left[\dfrac{\text{mol}}{\text{m}^3\,\text{Pa}}\right]$ | $\dfrac{\mathrm{d}\ln H_s^{cp}}{\mathrm{d}(1/T)}$ [K] | Reference | Type | Note |
|---|---|---|---|---|---|
| | $2.8\times10^{-1}$ | | Yaws (2003) | X | 258 |
| | $2.6\times10^{-1}$ | | Dupeux et al. (2022) | Q | 259 |
| | 1.4 | | Duchowicz et al. (2020) | Q | |
| | $2.9\times10^{-1}$ | | Zhang et al. (2010) | Q | 287, 288 |
| | $3.6\times10^{-1}$ | | Zhang et al. (2010) | Q | 287, 289 |
| | $9.5\times10^{-1}$ | | Zhang et al. (2010) | Q | 287, 290 |
| | $6.1\times10^{-1}$ | | Zhang et al. (2010) | Q | 287, 291 |
| | $2.9\times10^{-1}$ | | Hilal et al. (2008) | Q | |
| | $1.6\times10^{-1}$ | | Modarresi et al. (2007) | Q | 67 |
| | | 5100 | Kühne et al. (2005) | Q | |
| | $3.2\times10^{-1}$ | | Yaffe et al. (2003) | Q | 248, 249 |
| | $6.8\times10^{-1}$ | | Yao et al. (2002) | Q | 229 |
| | 1.1 | | English and Carroll (2001) | Q | 230, 274 |
| | $3.0\times10^{-1}$ | | Katritzky et al. (1998) | Q | |
| | $5.6\times10^{-1}$ | | Suzuki et al. (1992) | Q | 232 |
| | $2.8\times10^{-1}$ | | Meylan and Howard (1991) | Q | |
| | $2.7\times10^{-1}$ | | Nirmalakhandan and Speece (1988) | Q | |
| | | 3500 | Kühne et al. (2005) | ? | |
| | $2.8\times10^{-1}$ | | Yaws (1999) | ? | 21 |
| | $5.6\times10^{-1}$ | | Abraham et al. (1990) | ? | |
| ethyl benzoate $C_6H_5COOC_2H_5$ [93-89-0] MTZQAGJQAFMTAQ-UHFFFAOYSA-N | $1.7\times10^{-1}$ | 6300 | Brockbank (2013) | L | 1 |
| | $1.3\times10^{-1}$ | | Duchowicz et al. (2020) | V | 186 |
| | $9.7\times10^{-2}$ | | Mackay et al. (2006c) | V | |
| | $9.7\times10^{-2}$ | | Mackay et al. (1995) | V | |
| | $1.9\times10^{-1}$ | | Abraham et al. (1994a) | R | |
| | $1.4\times10^{-1}$ | | Yaws (2003) | X | 258 |
| | $1.5\times10^{-1}$ | | Dupeux et al. (2022) | Q | 259 |
| | $5.2\times10^{-1}$ | | Duchowicz et al. (2020) | Q | |
| | $3.1\times10^{-1}$ | | Raventos-Duran et al. (2010) | Q | 242, 243 |
| | $2.0\times10^{-1}$ | | Raventos-Duran et al. (2010) | Q | 244 |
| | $2.0\times10^{-1}$ | | Raventos-Duran et al. (2010) | Q | 245 |
| | $2.1\times10^{-1}$ | | Zhang et al. (2010) | Q | 287, 288 |
| | $2.1\times10^{-1}$ | | Zhang et al. (2010) | Q | 287, 289 |
| | $5.1\times10^{-1}$ | | Zhang et al. (2010) | Q | 287, 290 |
| | $4.8\times10^{-1}$ | | Zhang et al. (2010) | Q | 287, 291 |
| | $1.9\times10^{-1}$ | | Hilal et al. (2008) | Q | |
| | $1.6\times10^{-1}$ | | Modarresi et al. (2007) | Q | 67 |
| | $1.9\times10^{-1}$ | | Yaffe et al. (2003) | Q | 248, 249 |
| | $6.7\times10^{-1}$ | | English and Carroll (2001) | Q | 230, 231 |
| | $5.1\times10^{-1}$ | | Katritzky et al. (1998) | Q | |
| | $2.2\times10^{-1}$ | | Nirmalakhandan et al. (1997) | Q | |
| | $1.4\times10^{-1}$ | | Yaws (1999) | ? | 21 |
| | $1.9\times10^{-1}$ | | Abraham et al. (1990) | ? | |





Table A3.8: Esters (RCOOR) (. . . continued)

| Substance Formula (Trivial Name) [CAS Registry Number] InChIKey | $H_s^{cp}$ (at $T^{\ominus}$) $\left[\dfrac{\mathrm{mol}}{\mathrm{m}^3\,\mathrm{Pa}}\right]$ | $\dfrac{\mathrm{d}\ln H_s^{cp}}{\mathrm{d}(1/T)}$ [K] | Reference | Type | Note |
|---|---|---|---|---|---|
| 2-hydroxybenzoic acid methyl ester | 1.6 | 9500 | Brockbank (2013) | L | 1 |
| $C_8H_8O_3$ | $3.3\times10^{-1}$ | | Karl et al. (2008) | M | |
| (methyl salicylate) | $1.0\times10^{-1}$ | | Duchowicz et al. (2020) | V | 186 |
| [119-36-8] | $1.1\times10^{1}$ | | HSDB (2015) | V | |
| OSWPMRLSEDHDFF-UHFFFAOYSA-N | 7.8 | | Duchowicz et al. (2020) | Q | |
| | $1.8\times10^{1}$ | | Hilal et al. (2008) | Q | |
| | 9.3 | | Modarresi et al. (2007) | Q | 67 |
| | $1.0\times10^{-1}$ | | Bartelt-Hunt et al. (2008) | ? | 21 |
| | 1.0 | | Yaws (1999) | ? | 21, 38 |
| benzoic acid, 4-methyl-, methyl ester | $2.6\times10^{-1}$ | | Zhang et al. (2010) | Q | 287, 288 |
| $C_9H_{10}O_2$ | $3.9\times10^{-1}$ | | Zhang et al. (2010) | Q | 287, 289 |
| [99-75-2] | 1.7 | | Zhang et al. (2010) | Q | 287, 290 |
| QSSJZLPUHJDYKF-UHFFFAOYSA-N | $3.5\times10^{-1}$ | | Zhang et al. (2010) | Q | 287, 291 |
| acetylsalicylic acid | $7.6\times10^{3}$ | | Duchowicz et al. (2020) | V | 186 |
| $C_9H_8O_4$ | $1.5\times10^{4}$ | | Duchowicz et al. (2020) | Q | |
| (aspirin) | $1.5\times10^{5}$ | | Abraham et al. (2019) | Q | |
| [50-78-2] | $6.2\times10^{2}$ | | Raventos-Duran et al. (2010) | Q | 271, 243 |
| BSYNRYMUTXBXSQ-UHFFFAOYSA-N | $4.9\times10^{4}$ | | Raventos-Duran et al. (2010) | Q | 244 |
| | $7.8\times10^{3}$ | | Raventos-Duran et al. (2010) | Q | 245 |
| 1,4-benzenedicarboxylic acid, dimethyl ester | $7.4\times10^{-2}$ | | Duchowicz et al. (2020) | V | 186 |
| $C_{10}H_{10}O_4$ | $7.6\times10^{-2}$ | | HSDB (2015) | V | |
| [120-61-6] | $2.2\times10^{2}$ | | Duchowicz et al. (2020) | Q | |
| WOZVHXUHUFLZGK-UHFFFAOYSA-N | $4.4\times10^{1}$ | | Zhang et al. (2010) | Q | 287, 288 |
| | $1.3\times10^{2}$ | | Zhang et al. (2010) | Q | 287, 289 |
| | $5.3\times10^{1}$ | | Zhang et al. (2010) | Q | 287, 290 |
| | $9.2\times10^{1}$ | | Zhang et al. (2010) | Q | 287, 291 |
| | 2.4 | | Modarresi et al. (2007) | Q | 67 |
| butyl benzoate | $2.5\times10^{-1}$ | | Duchowicz et al. (2020) | V | 186 |
| $C_{11}H_{14}O_2$ | $1.0\times10^{-1}$ | | Dupeux et al. (2022) | Q | 259 |
| [136-60-7] | $5.7\times10^{-1}$ | | Duchowicz et al. (2020) | Q | |
| XSIFPSYPOVKYCO-UHFFFAOYSA-N | $2.0\times10^{-1}$ | | Raventos-Duran et al. (2010) | Q | 242, 243 |
| | $9.9\times10^{-2}$ | | Raventos-Duran et al. (2010) | Q | 244 |
| | $1.2\times10^{-1}$ | | Raventos-Duran et al. (2010) | Q | 245 |
| | $1.2\times10^{-1}$ | | Zhang et al. (2010) | Q | 287, 288 |
| | $1.0\times10^{-1}$ | | Zhang et al. (2010) | Q | 287, 289 |
| | $5.2\times10^{-1}$ | | Zhang et al. (2010) | Q | 287, 290 |
| | $3.2\times10^{-1}$ | | Zhang et al. (2010) | Q | 287, 291 |
| | $8.3\times10^{-2}$ | | Modarresi et al. (2007) | Q | 67 |



Table A3.8: Esters (RCOOR) (...continued)

| Substance / Formula / (Trivial Name) / [CAS Registry Number] / InChIKey | $H_s^{cp}$ (at $T^{\ominus}$) $\left[\dfrac{\mathrm{mol}}{\mathrm{m^3\,Pa}}\right]$ | $\dfrac{\mathrm{d}\ln H_s^{cp}}{\mathrm{d}(1/T)}$ [K] | Reference | Type | Note |
|---|---|---|---|---|---|
| diphenyl carbonate | $1.2\times10^{-1}$ | | HSDB (2015) | Q | 99 |
| $C_{13}H_{10}O_3$ | $1.2\times10^{-1}$ | | Zhang et al. (2010) | Q | 287, 288 |
| [102-09-0] | $1.6\times10^{1}$ | | Zhang et al. (2010) | Q | 287, 289 |
| ROORDVPLFPIABK-UHFFFAOYSA-N | $9.5\times10^{-1}$ | | Zhang et al. (2010) | Q | 287, 290 |
| | $1.2\times10^{2}$ | | Zhang et al. (2010) | Q | 287, 291 |
| benzyl benzoate | 1.8 | | Mackay et al. (2006c) | V | |
| $C_{14}H_{12}O_2$ | 1.8 | | Mackay et al. (1995) | V | |
| [120-51-4] | 1.7 | | Dupeux et al. (2022) | Q | 259 |
| SESFRYSPDFLNCH-UHFFFAOYSA-N | | | | | |
| *trans*-ethylcinnamate | 2.3 | | Duchowicz et al. (2020) | V | 186 |
| $C_{11}H_{12}O_2$ | 1.4 | | Abney (2021) | Q | 399 |
| [103-36-6] | $6.5\times10^{-1}$ | | Duchowicz et al. (2020) | Q | |
| KBEBGUQPQBELIU-CMDGGOBGSA-N | | | | | |
| dipropylene glycol dibenzoate | $7.2\times10^{2}$ | | Duchowicz et al. (2020) | V | 186 |
| $C_{20}H_{22}O_5$ | $9.4\times10^{2}$ | | Duchowicz et al. (2020) | Q | |
| [27138-31-4] | | | | | |
| CGLQOIMEUPORRI-UHFFFAOYSA-N | | | | | |
| dimethyl phthalate | $5.1\times10^{1}$ | 8000 | Brockbank (2013) | L | 1 |
| $C_{10}H_{10}O_4$ | $2.3\times10^{1}$ | | Chao et al. (2017) | M | |
| [131-11-3] | $5.0\times10^{1}$ | | Duchowicz et al. (2020) | V | 186 |
| NIQCNGHVCWTJSM-UHFFFAOYSA-N | $4.9\times10^{1}$ | | HSDB (2015) | V | |
| | $9.3\times10^{1}$ | | Mackay et al. (2006c) | V | |
| | $2.0\times10^{1}$ | | Saçan et al. (2005) | V | |
| | $1.0\times10^{2}$ | | Cousins and Mackay (2000) | V | |
| | $8.1\times10^{1}$ | | Staples et al. (1997) | V | |
| | $9.1\times10^{1}$ | | Lide and Frederikse (1995) | V | |
| | $9.1\times10^{1}$ | | Mackay et al. (1995) | V | |
| | $5.0\times10^{1}$ | | Hwang et al. (1992) | V | |
| | 9.0 | | Wolfe et al. (1980) | V | |
| | $8.8\times10^{1}$ | | Yaws (2003) | X | 237, 12 |
| | $2.9\times10^{1}$ | | Goldstein (1982) | X | 446 |
| | $3.0\times10^{1}$ | 5700 | Goldstein (1982) | X | 298 |
| | $2.3\times10^{1}$ | | McCarty (1980) | X | 368 |
| | $5.0\times10^{1}$ | | Ryan et al. (1988) | C | |
| | $2.2\times10^{2}$ | | Duchowicz et al. (2020) | Q | |
| | $4.9\times10^{1}$ | | Raventos-Duran et al. (2010) | Q | 242, 243 |
| | $3.9\times10^{2}$ | | Raventos-Duran et al. (2010) | Q | 244 |
| | $4.9\times10^{1}$ | | Raventos-Duran et al. (2010) | Q | 245 |
| | $8.2\times10^{1}$ | | Gharagheizi et al. (2010) | Q | 246 |
| | $1.7\times10^{2}$ | | Hilal et al. (2008) | Q | |
| | 9.6 | | Saçan et al. (2005) | Q | |
| | $5.4\times10^{1}$ | | Yaws (1999) | ? | 21, 12 |





Table A3.8: Esters (RCOOR) (...continued)

| Substance Formula (Trivial Name) [CAS Registry Number] InChIKey | $H_s^{cp}$ (at $T^\ominus$) $\left[\dfrac{\mathrm{mol}}{\mathrm{m^3\,Pa}}\right]$ | $\dfrac{\mathrm{d}\ln H_s^{cp}}{\mathrm{d}(1/T)}$ [K] | Reference | Type | Note |
|---|---|---|---|---|---|
| 1,4-cyclohexanedicarboxylic acid, dimethyl ester C$_{10}$H$_{16}$O$_4$ (dimethyl hexahydroterephthalate) [94-60-0] LNGAGQAGYITKCW-UHFFFAOYSA-N | $1.0\times10^2$ | | HSDB (2015) | V | |
| 1,3-benzenedicarboxylic acid, dimethyl ester C$_{10}$H$_{10}$O$_4$ (dimethyl isophthalate) [1459-93-4] VNGOYPQMJFJDLV-UHFFFAOYSA-N | $1.6\times10^2$ | | HSDB (2015) | Q | 99 |
| diethyl phthalate C$_{12}$H$_{14}$O$_4$ [84-66-2] FLKPEMZONWLCSK-UHFFFAOYSA-N | $6.0\times10^1$ | 10000 | Brockbank (2013) | L | 1 |
| | $2.2\times10^1$ | | Chao et al. (2017) | M | |
| | $1.6\times10^1$ | | Duchowicz et al. (2020) | V | 186 |
| | $1.6\times10^1$ | | HSDB (2015) | V | |
| | $2.2\times10^1$ | | Mackay et al. (2006c) | V | |
| | $4.1\times10^1$ | | Cousins and Mackay (2000) | V | |
| | $3.7\times10^1$ | | Staples et al. (1997) | V | |
| | $2.1\times10^1$ | | Lide and Frederikse (1995) | V | |
| | $1.0\times10^2$ | | Mackay et al. (1995) | V | |
| | $4.9\times10^2$ | | Wolfe et al. (1980) | V | |
| | $6.9\times10^1$ | | Yaws (2003) | X | 237 |
| | $1.2\times10^1$ | | Goldstein (1982) | X | 446 |
| | $1.2\times10^1$ | 5600 | Goldstein (1982) | X | 298 |
| | $2.1\times10^{-1}$ | | Ryan et al. (1988) | C | |
| | $5.8\times10^{-1}$ | | Petrasek et al. (1983) | C | |
| | $3.4\times10^1$ | | Duchowicz et al. (2020) | Q | |
| | $1.5\times10^1$ | | Gharagheizi et al. (2012) | Q | |
| | $2.5\times10^1$ | | Zhang et al. (2010) | Q | 287, 288 |
| | $1.5\times10^2$ | | Zhang et al. (2010) | Q | 287, 289 |
| | $2.7\times10^2$ | | Zhang et al. (2010) | Q | 287, 290 |
| | $5.6\times10^1$ | | Zhang et al. (2010) | Q | 287, 291 |
| | $6.5\times10^1$ | | Gharagheizi et al. (2010) | Q | 246 |
| | $7.7\times10^1$ | | Hilal et al. (2008) | Q | |
| | | 12000 | Kühne et al. (2005) | Q | |
| | 5.8 | | Saçan et al. (2005) | Q | |
| | $2.1\times10^1$ | | Yao et al. (2002) | Q | 229 |
| | $1.6\times10^1$ | | Bartelt-Hunt et al. (2008) | ? | 21 |
| | | 12000 | Kühne et al. (2005) | ? | |
| | $6.9\times10^1$ | | Yaws (1999) | ? | 21 |





Table A3.8: Esters (RCOOR) (...continued)

| Substance<br>Formula<br>(Trivial Name)<br>[CAS Registry Number]<br>InChIKey | $H_s^{cp}$<br>(at $T^{\ominus}$)<br>$\left[\dfrac{\text{mol}}{\text{m}^3\,\text{Pa}}\right]$ | $\dfrac{\text{d}\ln H_s^{cp}}{\text{d}(1/T)}$<br><br>[K] | Reference | Type | Note |
|---|---|---|---|---|---|
| dipropyl phthalate | $2.2\times10^1$ | | Brockbank (2013) | L | |
| $C_{14}H_{18}O_4$ | $2.4\times10^1$ | | Duchowicz et al. (2020) | V | 186 |
| [131-16-8] | $1.8\times10^1$ | | Cousins and Mackay (2000) | V | 516 |
| MQHNKCZKNAJROC-UHFFFAOYSA-N | 3.3 | | Cousins and Mackay (2000) | V | 516 |
| | $3.2\times10^1$ | | Staples et al. (1997) | V | |
| | $4.1\times10^1$ | | Duchowicz et al. (2020) | Q | |
| | $1.6\times10^1$ | | Raventos-Duran et al. (2010) | Q | 271, 243 |
| | $7.8\times10^1$ | | Raventos-Duran et al. (2010) | Q | 244 |
| | $1.6\times10^1$ | | Raventos-Duran et al. (2010) | Q | 245 |
| | $2.4\times10^1$ | | Saçan et al. (2005) | Q | |
| diallyl phthalate | $3.5\times10^1$ | | Saçan et al. (2005) | V | |
| $C_{14}H_{14}O_4$ | $2.3\times10^1$ | | Cousins and Mackay (2000) | V | |
| [131-17-9] | $3.5\times10^1$ | | Staples et al. (1997) | V | |
| QUDWYFHPNIMBFC-UHFFFAOYSA-N | $2.5\times10^1$ | | HSDB (2015) | Q | 99 |
| | $1.7\times10^1$ | | Saçan et al. (2005) | Q | |
| bis(2-methoxyethyl) phthalate | $2.3\times10^1$ | | Fishbein and Albro (1972) | V | 12 |
| $C_{14}H_{18}O_5$ | $3.5\times10^7$ | | HSDB (2015) | Q | 99 |
| [117-82-8] | | | | | |
| HSUIVCLOAAJSRE-UHFFFAOYSA-N | | | | | |
| dibutyl phthalate | $1.1\times10^1$ | 12000 | Brockbank (2013) | L | 1, 517 |
| $C_{16}H_{22}O_4$ | 9.3 | | Lee et al. (2012) | M | |
| [84-74-2] | 5.5 | | Atlas et al. (1983) | M | 72 |
| DOIRQSBPFJWKBE-UHFFFAOYSA-N | $2.2\times10^1$ | | Mackay et al. (2006c) | V | |
| | $2.7\times10^1$ | | Saçan et al. (2005) | V | |
| | 7.5 | | Cousins and Mackay (2000) | V | |
| | $1.1\times10^1$ | | Staples et al. (1997) | V | |
| | $2.2\times10^1$ | | Lide and Frederikse (1995) | V | |
| | $2.0\times10^1$ | | Mackay et al. (1995) | V | |
| | $2.6\times10^2$ | | Hwang et al. (1992) | V | |
| | 7.6 | | Wolfe et al. (1980) | V | |
| | $5.6\times10^1$ | | Yaws (2003) | X | 237, 12 |
| | $1.6\times10^{-1}$ | | McCarty (1980) | X | 368 |
| | $3.4\times10^1$ | | Ryan et al. (1988) | C | |
| | $5.6\times10^1$ | | Gharagheizi et al. (2010) | Q | 246 |
| | $2.9\times10^1$ | | Hilal et al. (2008) | Q | |
| | 1.1 | | Modarresi et al. (2007) | Q | 67 |
| | | 14000 | Kühne et al. (2005) | Q | |
| | $3.7\times10^1$ | | Saçan et al. (2005) | Q | |
| | | 13000 | Kühne et al. (2005) | ? | |
| | $2.7\times10^1$ | | Yaws (1999) | ? | 21, 12 |
| diisobutyl phthalate | 3.5 | | HSDB (2015) | V | |
| $C_{16}H_{22}O_4$ | 7.5 | | Cousins and Mackay (2000) | V | |
| [84-69-5] | $5.4\times10^1$ | | Staples et al. (1997) | V | |
| MGWAVDBGNNKXQV-UHFFFAOYSA-N | $3.1\times10^1$ | | Saçan et al. (2005) | Q | |



Table A3.8: Esters (RCOOR) (...continued)

| Substance Formula (Trivial Name) [CAS Registry Number] InChIKey | $H_s^{cp}$ (at $T^\ominus$) $\left[\dfrac{\mathrm{mol}}{\mathrm{m^3\,Pa}}\right]$ | $\dfrac{\mathrm{d}\ln H_s^{cp}}{\mathrm{d}(1/T)}$ [K] | Reference | Type | Note |
|---|---|---|---|---|---|
| 1,2-benzenedicarboxylic acid, butyl cyclohexyl ester $C_{18}H_{24}O_4$ (butyl cyclohexyl phthalate) [84-64-0] BHKLONWXRPJNAE-UHFFFAOYSA-N | $1.0\times10^1$ | | HSDB (2015) | Q | 99 |
| butyl glycolyl butyl phthalate $C_{18}H_{24}O_6$ [85-70-1] GOJCZVPJCKEBQV-UHFFFAOYSA-N | $4.7\times10^2$ | | HSDB (2015) | Q | 99 |
| diamyl phthalate $C_{18}H_{26}O_4$ [131-18-0] IPKKHRVROFYTEK-UHFFFAOYSA-N | $1.1\times10^1$ | | HSDB (2015) | Q | 99 |
| butyl benzyl phthalate $C_{19}H_{20}O_4$ [85-68-7] IRIAEXORFWYRCZ-UHFFFAOYSA-N | $1.0\times10^2$ | | Lee et al. (2012) | M | |
| | 7.8 | | Duchowicz et al. (2020) | V | 186 |
| | 7.6 | | HSDB (2015) | V | |
| | 7.5 | | Mackay et al. (2006c) | V | |
| | $1.9\times10^1$ | | Saçan et al. (2005) | V | |
| | 4.9 | | Cousins and Mackay (2000) | V | |
| | $1.3\times10^1$ | | Staples et al. (1997) | V | |
| | 7.8 | | Mackay et al. (1995) | V | |
| | 9.6 | | Ryan et al. (1988) | C | |
| | $2.9\times10^2$ | | Duchowicz et al. (2020) | Q | |
| | $3.2\times10^1$ | | Saçan et al. (2005) | Q | |
| | >9.9 | | Petrasek et al. (1983) | E | |
| dihexyl phthalate $C_{20}H_{30}O_4$ [84-75-3] KCXZNSGUUQJJTR-UHFFFAOYSA-N | 8.3 | | Brockbank (2013) | L | |
| | $3.8\times10^{-1}$ | | Duchowicz et al. (2020) | V | 186 |
| | $3.8\times10^{-1}$ | | HSDB (2015) | V | |
| | 1.4 | | Cousins and Mackay (2000) | V | |
| | $2.2\times10^{-1}$ | | Staples et al. (1997) | V | |
| | $5.5\times10^1$ | | Duchowicz et al. (2020) | Q | |
| | $1.6\times10^1$ | | Saçan et al. (2005) | Q | |
| butyl 2-ethylhexyl phthalate $C_{20}H_{30}O_4$ [85-69-8] AVOLBYOSCILFLL-UHFFFAOYSA-N | 2.1 | | Cousins and Mackay (2000) | V | |
| | $2.5\times10^1$ | | Staples et al. (1997) | V | |
| | 4.7 | | HSDB (2015) | Q | 99 |
| | $6.9\times10^1$ | | Saçan et al. (2005) | Q | |
| diphenyl terephthalate $C_{20}H_{14}O_4$ [1539-04-4] HPGJOUYGWKFYQW-UHFFFAOYSA-N | $3.2\times10^2$ | | Zhang et al. (2010) | Q | 287, 288 |
| | $4.3\times10^4$ | | Zhang et al. (2010) | Q | 287, 289 |
| | $2.7\times10^4$ | | Zhang et al. (2010) | Q | 287, 290 |
| | $7.7\times10^4$ | | Zhang et al. (2010) | Q | 287, 291 |



Table A3.8: Esters (RCOOR) (. . . continued)

| Substance Formula (Trivial Name) [CAS Registry Number] InChIKey | $H_s^{cp}$ (at $T^{\ominus}$) $\left[\dfrac{\mathrm{mol}}{\mathrm{m^3\,Pa}}\right]$ | $\dfrac{\mathrm{d}\ln H_s^{cp}}{\mathrm{d}(1/T)}$ [K] | Reference | Type | Note |
|---|---|---|---|---|---|
| dicyclohexyl phthalate $C_{20}H_{26}O_4$ [84-61-7] VOWAEIGWURALJQ-UHFFFAOYSA-N | $9.9\times10^1$ $9.9\times10^1$ $4.8\times10^2$ | | Duchowicz et al. (2020) HSDB (2015) Duchowicz et al. (2020) | V V Q | 186 |
| bis(2-butoxyethyl) phthalate $C_{20}H_{30}O_6$ [117-83-9] CMCJNODIWQEOAI-UHFFFAOYSA-N | $4.9\times10^6$ | | HSDB (2015) | Q | 99 |
| diheptyl phthalate $C_{22}H_{34}O_4$ [3648-21-3] JQCXWCOOWVGKMT-UHFFFAOYSA-N | $5.9\times10^{-1}$ 2.8 $8.9\times10^{-1}$ | | Cousins and Mackay (2000) HSDB (2015) Saçan et al. (2005) | V Q Q | 99 |
| dioctyl phthalate $C_{24}H_{38}O_4$ [117-84-0] MQIUGAXCHLFZKX-UHFFFAOYSA-N | 3.8 3.8 $9.6\times10^{-2}$ $2.5\times10^{-1}$ $9.6\times10^{-2}$ 1.8 1.8 $8.9\times10^1$ $3.4\times10^1$ $6.2\times10^1$ $7.2\times10^1$ 6.4 $>9.9$ $7.6\times10^1$ | | Duchowicz et al. (2020) HSDB (2015) Mackay et al. (2006c) Cousins and Mackay (2000) Staples et al. (1997) Mackay et al. (1995) Wolfe et al. (1980) Yaws (2003) Ryan et al. (1988) Duchowicz et al. (2020) Gharagheizi et al. (2010) Saçan et al. (2005) Petrasek et al. (1983) Yaws (1999) | V V V V V V V X C Q Q Q E ? | 186 237, 79 246 21, 79 |
| diisooctyl phthalate $C_{24}H_{38}O_4$ [27554-26-3] IJFPVINAQGWBRJ-UHFFFAOYSA-N | $3.1\times10^{-1}$ $3.2\times10^{-1}$ $2.5\times10^{-1}$ 9.5 $1.5\times10^2$ | | Duchowicz et al. (2020) HSDB (2015) Cousins and Mackay (2000) Duchowicz et al. (2020) Saçan et al. (2005) | V V V Q Q | 186 |
| decyl hexyl phthalate $C_{24}H_{38}O_4$ [25724-58-7] OMQBXAQAHHFSST-UHFFFAOYSA-N | $1.6\times10^2$ | | Saçan et al. (2005) | Q | |
| bis(2-ethylhexyl)-phthalate $C_{24}H_{38}O_4$ (DEHP) [117-81-7] BJQHLKABXJIVAM-UHFFFAOYSA-N | $3.7\times10^1$ $3.7\times10^1$ $5.8\times10^{-1}$ $1.4\times10^1$ $2.5\times10^{-1}$ $5.8\times10^{-1}$ $3.7\times10^1$ $6.7\times10^{-1}$ $5.4\times10^{-2}$ | | Duchowicz et al. (2020) HSDB (2015) Mackay et al. (2006c) Saçan et al. (2005) Cousins and Mackay (2000) Staples et al. (1997) Mackay et al. (1995) Meylan and Howard (1991) Riederer (1990) | V V V V V V V V V | 186 |





Table A3.8: Esters (RCOOR) (...continued)

| Substance Formula (Trivial Name) [CAS Registry Number] InChIKey | $H_s^{cp}$ (at $T^\ominus$) $\left[\dfrac{\text{mol}}{\text{m}^3\,\text{Pa}}\right]$ | $\dfrac{\text{d}\ln H_s^{cp}}{\text{d}(1/T)}$ [K] | Reference | Type | Note |
|---|---|---|---|---|---|
| | $2.2\times10^1$ | | Wolfe et al. (1980) | V | |
| | $3.4\times10^1$ | | Ryan et al. (1988) | C | |
| | $8.2\times10^1$ | | Petrasek et al. (1983) | C | |
| | $9.5$ | | Duchowicz et al. (2020) | Q | |
| | $2.5\times10^1$ | | Saçan et al. (2005) | Q | |
| | $8.4\times10^{-1}$ | | Meylan and Howard (1991) | Q | |
| | $3.7\times10^1$ | | Bartelt-Hunt et al. (2008) | ? | 21 |
| bis(2-ethylhexyl) terephthalate $C_{24}H_{38}O_4$ [6422-86-2] RWPICVVBGZBXNA-UHFFFAOYSA-N | $9.9\times10^{-1}$ | | HSDB (2015) | Q | 99 |
| dinonyl phthalate $C_{26}H_{42}O_4$ [84-76-4] DROMNWUQASBTFM-UHFFFAOYSA-N | $1.1\times10^{-1}$ $7.0\times10^{-1}$ $3.0\times10^1$ | | Cousins and Mackay (2000) HSDB (2015) Saçan et al. (2005) | V Q Q | 99 |
| diisononyl phthalate $C_{26}H_{42}O_4$ [28553-12-0] HBGGXOJOCNVPFY-UHFFFAOYSA-N | $6.6$ $6.6$ $1.1\times10^{-1}$ $9.9$ $3.3\times10^1$ | | Duchowicz et al. (2020) HSDB (2015) Cousins and Mackay (2000) Duchowicz et al. (2020) Saçan et al. (2005) | V V V Q Q | 186 |
| didecyl phthalate $C_{28}H_{46}O_4$ [84-77-5] PGIBJVOPLXHHGS-UHFFFAOYSA-N | $4.6\times10^{-2}$ $3.5\times10^{-1}$ | | Cousins and Mackay (2000) HSDB (2015) | V Q | 99 |
| diisodecyl phthalate $C_{28}H_{46}O_4$ [26761-40-0] ZVFDTKUVRCTHQE-UHFFFAOYSA-N | $8.9$ $9.0$ $3.8\times10^1$ $4.6\times10^{-2}$ $1.1\times10^1$ $1.0\times10^1$ $1.3\times10^1$ $2.4\times10^1$ $8.9$ | | Duchowicz et al. (2020) HSDB (2015) Saçan et al. (2005) Cousins and Mackay (2000) Yaws (2003) Duchowicz et al. (2020) Gharagheizi et al. (2010) Saçan et al. (2005) Yaws (1999) | V V V V X Q Q Q ? | 186 237, 79 246 21, 79 |
| diundecyl phthalate $C_{30}H_{50}O_4$ [3648-20-2] QQVHEQUEHCEAKS-UHFFFAOYSA-N | $3.3\times10^1$ $2.0\times10^{-2}$ $1.8\times10^{-1}$ $1.4\times10^1$ | | Saçan et al. (2005) Cousins and Mackay (2000) HSDB (2015) Saçan et al. (2005) | V V Q Q | 99 |
| ditridecyl phthalate $C_{34}H_{58}O_4$ [119-06-2] YCZJVRCZIPDYHH-UHFFFAOYSA-N | $3.6\times10^{-3}$ $4.5\times10^{-2}$ $7.9\times10^1$ | | Cousins and Mackay (2000) HSDB (2015) Saçan et al. (2005) | V Q Q | 99 |



Table A3.8: Esters (RCOOR) (...continued)

| Substance<br>Formula<br>(Trivial Name)<br>[CAS Registry Number]<br>InChIKey | $H_s^{cp}$<br>(at $T^\ominus$)<br>$\left[\dfrac{\text{mol}}{\text{m}^3\,\text{Pa}}\right]$ | $\dfrac{\mathrm{d}\ln H_s^{cp}}{\mathrm{d}(1/T)}$<br><br>[K] | Reference | Type | Note |
|---|---|---|---|---|---|
| ethanedioic acid, dimethyl ester<br>$C_4H_6O_4$<br>(dimethyl oxalate)<br>[553-90-2]<br>LOMVENUNSWAXEN-UHFFFAOYSA-N | $3.4$<br>$2.6\times10^1$<br>$6.2\times10^{-1}$<br>$1.6\times10^1$<br>$3.9$<br>$6.9$<br>$4.6\times10^{-1}$ | | Duchowicz et al. (2020)<br>Duchowicz et al. (2020)<br>Raventos-Duran et al. (2010)<br>Raventos-Duran et al. (2010)<br>Raventos-Duran et al. (2010)<br>Hilal et al. (2008)<br>Modarresi et al. (2007) | V<br>Q<br>Q<br>Q<br>Q<br>Q<br>Q | 186<br><br>242, 243<br>244<br>245<br><br>67 |
| ethanedioic acid, diethyl ester<br>$C_6H_{10}O_4$<br>[95-92-1]<br>WYACBZDAHNBPPB-UHFFFAOYSA-N | $4.5$<br>$6.4$ | | Duchowicz et al. (2020)<br>Duchowicz et al. (2020) | V<br>Q | 186 |
| propanedioic acid, dimethyl ester<br>$C_5H_8O_4$<br>(dimethyl malonate)<br>[108-59-8]<br>BEPAFCGSDWSTEL-UHFFFAOYSA-N | $3.8\times10^1$ | 11000 | Katrib et al. (2003) | M | |
| propanedioic acid, diethyl ester<br>$C_7H_{12}O_4$<br>(diethyl malonate)<br>[105-53-3]<br>IYXGSMUGOJNHAZ-UHFFFAOYSA-N | $3.9$<br>$4.7$<br>$7.9$<br>$2.0$<br>$4.9$<br>$1.2\times10^1$<br>$3.9$<br>$5.0\times10^{-1}$<br><br>$4.2$<br><br>$4.1$ | 7500<br><br><br><br><br><br><br><br>5900<br><br>6400 | Brockbank (2013)<br>Duchowicz et al. (2020)<br>Duchowicz et al. (2020)<br>Raventos-Duran et al. (2010)<br>Raventos-Duran et al. (2010)<br>Raventos-Duran et al. (2010)<br>Hilal et al. (2008)<br>Modarresi et al. (2007)<br>Kühne et al. (2005)<br>Bartelt-Hunt et al. (2008)<br>Kühne et al. (2005)<br>Yaws (1999) | L<br>V<br>Q<br>Q<br>Q<br>Q<br>Q<br>Q<br>Q<br>?<br>?<br>? | 1<br>186<br><br>242, 243<br>244<br>245<br><br>67<br><br>21<br><br>21, 14 |
| butanedioic acid, dimethyl ester<br>$C_6H_{10}O_4$<br>(dimethyl succinate)<br>[106-65-0]<br>MUXOBHXGJLMRAB-UHFFFAOYSA-N | $3.0\times10^1$<br>$1.5\times10^2$ | 8500<br><br>7100<br>7000 | Katrib et al. (2003)<br>HSDB (2015)<br>Kühne et al. (2005)<br>Kühne et al. (2005) | M<br>Q<br>Q<br>? | <br>99 |
| diethyl succinate<br>$C_8H_{14}O_4$<br>[123-25-1]<br>DKMROQRQHGEIOW-UHFFFAOYSA-N | $1.9\times10^1$<br>$9.3$<br>$3.9\times10^1$<br>$4.9$<br>$9.9$<br>$4.0$<br>$1.3$ | | Duchowicz et al. (2020)<br>Duchowicz et al. (2020)<br>Raventos-Duran et al. (2010)<br>Raventos-Duran et al. (2010)<br>Raventos-Duran et al. (2010)<br>Hilal et al. (2008)<br>Modarresi et al. (2007) | V<br>Q<br>Q<br>Q<br>Q<br>Q<br>Q | 186<br><br>242, 243<br>244<br>245<br><br>67 |





Table A3.8: Esters (RCOOR) (...continued)

| Substance<br>Formula<br>(Trivial Name)<br>[CAS Registry Number]<br>InChIKey | $H_s^{cp}$<br>(at $T^\ominus$)<br>$\left[\dfrac{\mathrm{mol}}{\mathrm{m}^3\,\mathrm{Pa}}\right]$ | $\dfrac{\mathrm{d}\ln H_s^{cp}}{\mathrm{d}(1/T)}$<br><br>[K] | Reference | Type | Note |
|---|---|---|---|---|---|
| ($Z$)-2-butenedioic acid dimethyl ester | $1.4\times10^1$ | | Duchowicz et al. (2020) | V | 186 |
| $C_6H_8O_4$ | $5.2\times10^1$ | | Duchowicz et al. (2020) | Q | |
| (dimethyl maleate) | $1.2\times10^1$ | | Raventos-Duran et al. (2010) | Q | 271, 243 |
| [624-48-6] | $3.9\times10^1$ | | Raventos-Duran et al. (2010) | Q | 244 |
| LDCRTTXIJACKKU-ARJAWSKDSA-N | $7.8\times10^1$ | | Raventos-Duran et al. (2010) | Q | 245 |
| | $2.3\times10^1$ | | Hilal et al. (2008) | Q | |
| | 1.6 | | Modarresi et al. (2007) | Q | 67 |
| ($E$)-2-butenedioic acid dimethyl ester | $1.4\times10^1$ | | HSDB (2015) | Q | 99 |
| $C_6H_8O_4$ | | | | | |
| (dimethyl fumarate) | | | | | |
| [624-49-7] | | | | | |
| LDCRTTXIJACKKU-ONEGZZNKSA-N | | | | | |
| pentanedioic acid, dimethyl ester | $1.5\times10^1$ | | Duchowicz et al. (2020) | V | 186 |
| $C_7H_{12}O_4$ | $6.9\times10^1$ | | Duchowicz et al. (2020) | Q | |
| [1119-40-0] | | | | | |
| XTDYIOOONNVFMA-UHFFFAOYSA-N | | | | | |
| diethyl glutarate | $1.7\times10^1$ | | Ebert et al. (2023) | ? | 316 |
| $C_9H_{16}O_4$ | | | | | |
| [818-38-2] | | | | | |
| OUWSNHWQZPEFEX-UHFFFAOYSA-N | | | | | |
| diethyl pimelate | $2.2\times10^1$ | | Duchowicz et al. (2020) | V | 186 |
| $C_{11}H_{20}O_4$ | $1.3\times10^1$ | | Duchowicz et al. (2020) | Q | |
| [2050-20-6] | 1.5 | | Hilal et al. (2008) | Q | |
| LKKOGZVQGQUVHF-UHFFFAOYSA-N | $8.6\times10^{-1}$ | | Modarresi et al. (2007) | Q | 67 |
| | $2.2\times10^1$ | | Bartelt-Hunt et al. (2008) | ? | 21 |
| 1,3-benzenedicarboxylic acid, diethyl ester | $2.5\times10^1$ | | Zhang et al. (2010) | Q | 287, 288 |
| $C_{12}H_{14}O_4$ | $1.9\times10^1$ | | Zhang et al. (2010) | Q | 287, 289 |
| [636-53-3] | $2.9\times10^7$ | | Zhang et al. (2010) | Q | 287, 290 |
| JLVWYWVLMFVCDI-UHFFFAOYSA-N | $5.6\times10^1$ | | Zhang et al. (2010) | Q | 287, 291 |
| dibutyl maleate | $2.6\times10^1$ | | Duchowicz et al. (2020) | V | 186 |
| $C_{12}H_{20}O_4$ | $1.5\times10^1$ | | Duchowicz et al. (2020) | Q | |
| [105-76-0] | 2.5 | | Raventos-Duran et al. (2010) | Q | 271, 243 |
| JBSLOWBPDRZSMB-FPLPWBNLSA-N | 4.9 | | Raventos-Duran et al. (2010) | Q | 244 |
| | $1.2\times10^1$ | | Raventos-Duran et al. (2010) | Q | 245 |
| dimethyl sebacate | $1.8\times10^1$ | | Ebert et al. (2023) | ? | 316 |
| $C_{12}H_{22}O_4$ | | | | | |
| (decanedioic acid, dimethyl ester) | | | | | |
| [106-79-6] | | | | | |
| ALOUNLDAKADEEB-UHFFFAOYSA-N | | | | | |



Table A3.8: Esters (RCOOR) (...continued)

| Substance<br>Formula<br>(Trivial Name)<br>[CAS Registry Number]<br>InChIKey | $H_s^{cp}$<br>(at $T^{\ominus}$)<br>$\left[\dfrac{\text{mol}}{\text{m}^3\,\text{Pa}}\right]$ | $\dfrac{\text{d}\ln H_s^{cp}}{\text{d}(1/T)}$<br><br>[K] | Reference | Type | Note |
|---|---|---|---|---|---|
| 2,6-naphthalenedicarboxylic acid,<br>dimethyl ester | $4.5\times10^2$ | | Zhang et al. (2010) | Q | 287, 288 |
| $C_{14}H_{12}O_4$ | $2.5\times10^3$ | | Zhang et al. (2010) | Q | 287, 289 |
| [840-65-3] | $2.6\times10^7$ | | Zhang et al. (2010) | Q | 287, 290 |
| GYUVMLBYMPKZAZ-UHFFFAOYSA-N | $1.3\times10^3$ | | Zhang et al. (2010) | Q | 287, 291 |
| diisooctyl adipate | $1.9\times10^{-1}$ | | HSDB (2015) | Q | 99 |
| $C_{22}H_{42}O_4$ | | | | | |
| [1330-86-5] | | | | | |
| CJFLBOQMPJCWLR-UHFFFAOYSA-N | | | | | |
| di-(2-ethylhexyl)-adipate | $2.3\times10^1$ | | Felder et al. (1986) | M | 87 |
| $C_{22}H_{42}O_4$ | $4.3\times10^{-1}$ | | Hilal et al. (2008) | Q | |
| [103-23-1] | $4.0\times10^{-1}$ | | Modarresi et al. (2007) | Q | 67 |
| SAOKZLXYCUGLFA-UHFFFAOYSA-N | | | | | |
| peroxybenzoic acid, *tert*-butyl ester | $4.7\times10^{-2}$ | | HSDB (2015) | Q | 99 |
| $C_{11}H_{14}O_3$ | $4.7\times10^{-2}$ | | Zhang et al. (2010) | Q | 287, 288 |
| [614-45-9] | $1.8\times10^{-1}$ | | Zhang et al. (2010) | Q | 287, 289 |
| GJBRNHKUVLOCEB-UHFFFAOYSA-N | 8.2 | | Zhang et al. (2010) | Q | 287, 290 |
| | 5.4 | | Zhang et al. (2010) | Q | 287, 291 |
| neodecaneperoxoic acid,<br>1,1-dimethylethyl ester | $9.9\times10^{-4}$ | | Zhang et al. (2010) | Q | 287, 288 |
| $C_{14}H_{28}O_3$ | $4.7\times10^{-3}$ | | Zhang et al. (2010) | Q | 287, 289 |
| [26748-41-4] | $1.2\times10^{-1}$ | | Zhang et al. (2010) | Q | 287, 290 |
| NMOALOSNPWTWRH-UHFFFAOYSA-N | $2.3\times10^{-2}$ | | Zhang et al. (2010) | Q | 287, 291 |
| neoheptaneperoxoic acid,<br>1-methyl-1-phenylethyl ester | $3.8\times10^{-2}$ | | Zhang et al. (2010) | Q | 287, 288 |
| $C_{16}H_{24}O_3$ | $9.0\times10^{-2}$ | | Zhang et al. (2010) | Q | 287, 289 |
| [130097-36-8] | 2.5 | | Zhang et al. (2010) | Q | 287, 290 |
| WFAUFYAGXAXBEG-UHFFFAOYSA-N | 1.3 | | Zhang et al. (2010) | Q | 287, 291 |
| hydroxypropyl acrylate | $5.8\times10^3$ | | HSDB (2015) | Q | 99 |
| $C_6H_{10}O_3$ | | | | | |
| [25584-83-2] | | | | | |
| AYEFIAVHMUFQPZ-UHFFFAOYSA-N | | | | | |
| 2-hydroxyethyl methacrylate | $4.6\times10^1$ | | Duchowicz et al. (2020) | V | 186 |
| $C_6H_{10}O_3$ | $6.6\times10^1$ | | Duchowicz et al. (2020) | Q | |
| [868-77-9] | $2.1\times10^3$ | | HSDB (2015) | Q | 99 |
| WOBHKFSMXKNTIM-UHFFFAOYSA-N | | | | | |
| 2-hydroxypropyl acrylate | $1.6\times10^3$ | | HSDB (2015) | Q | 99 |
| $C_6H_{10}O_3$ | | | | | |
| [999-61-1] | | | | | |
| GWZMWHWAWHPNHN-UHFFFAOYSA-N | | | | | |

Table A3.8: Esters (RCOOR) (... continued)

| Substance<br>Formula<br>(Trivial Name)<br>[CAS Registry Number]<br>InChIKey | $H_s^{cp}$ (at $T^\ominus$) $\left[\frac{\text{mol}}{\text{m}^3\,\text{Pa}}\right]$ | $\frac{d\ln H_s^{cp}}{d(1/T)}$ [K] | Reference | Type | Note |
|---|---|---|---|---|---|
| hexanedioic acid, dimethyl ester<br>$C_8H_{14}O_4$<br>(dimethyl adipate)<br>[627-93-0]<br>UDSFAEKRVUSQDD-UHFFFAOYSA-N | 4.3 | | Duchowicz et al. (2020) | V | 186 |
| | $8.1\times10^{1}$ | | Duchowicz et al. (2020) | Q | |
| | $1.0\times10^{1}$ | | HSDB (2015) | Q | 99 |
| | 6.0 | | Bartelt-Hunt et al. (2008) | ? | 21 |
| methyl 4-hydroxybenzoate<br>$C_8H_8O_3$<br>(methylparaben)<br>[99-76-3]<br>LXCFILQKKLGQFO-UHFFFAOYSA-N | $4.5\times10^{3}$ | | HSDB (2015) | Q | 99 |
| diethyl fumarate<br>$C_8H_{12}O_4$<br>[623-91-6]<br>IEPRKVQEAMIZSS-AATRIKPKSA-N | $4.1\times10^{2}$ | | HSDB (2015) | Q | 99 |
| diethyl adipate<br>$C_{10}H_{18}O_4$<br>[141-28-6]<br>VIZORQUEIQEFRT-UHFFFAOYSA-N | 2.7 | | Duchowicz et al. (2020) | V | 186 |
| | 2.7 | | HSDB (2015) | V | |
| | $1.2\times10^{1}$ | | Duchowicz et al. (2020) | Q | |
| | 3.7 | | Bartelt-Hunt et al. (2008) | ? | 21 |
| propyl 4-hydroxybenzoate<br>$C_{10}H_{12}O_3$<br>(propylparaben)<br>[94-13-3]<br>QELSKZZBTMNZEB-UHFFFAOYSA-N | $1.5\times10^{3}$ | | HSDB (2015) | Q | 99 |
| diethylene glycol diacrylate<br>$C_{10}H_{14}O_5$<br>[4074-88-8]<br>LEJBBGNFPAFPKQ-UHFFFAOYSA-N | $1.0\times10^{4}$ | | HSDB (2015) | Q | 99 |
| (2,2-dimethyl-3-prop-2-enoyloxypropyl) prop-2-enoate<br>$C_{11}H_{16}O_4$<br>(2,2-dimethyltrimethylene acrylate)<br>[2223-82-7]<br>MXFQRSUWYYSPOC-UHFFFAOYSA-N | $2.7\times10^{1}$ | | HSDB (2015) | Q | 447 |
| methyl jasmonate<br>$C_{13}H_{20}O_3$<br>[1211-29-6]<br>GEWDNTWNSAZUDX-NNOMMRTBSA-N | $5.0\times10^{1}$ | | Karl et al. (2008) | M | |
| | $7.0\times10^{2}$ | | HSDB (2015) | Q | 99 |
| cinoxate<br>$C_{14}H_{18}O_4$<br>[104-28-9]<br>CMDKPGRTAQVGFQ-RMKNXTFCSA-N | $1.9\times10^{3}$ | | HSDB (2015) | Q | 447 |



Table A3.8: Esters (RCOOR) (... continued)

| Substance Formula (Trivial Name) [CAS Registry Number] InChIKey | $H_s^{cp}$ (at $T^{\ominus}$) $\left[\dfrac{\mathrm{mol}}{\mathrm{m^3\,Pa}}\right]$ | $\dfrac{\mathrm{d}\ln H_s^{cp}}{\mathrm{d}(1/T)}$ [K] | Reference | Type | Note |
|---|---|---|---|---|---|
| trimethylolpropane triacrylate $C_{15}H_{20}O_6$ [15625-89-5] DAKWPKUUDNSNPN-UHFFFAOYSA-N | $1.6{\times}10^4$ | | HSDB (2015) | Q | 447 |
| benzyl cinnamate $C_{16}H_{14}O_2$ [103-41-3] NGHOLYJTSCBCGC-VAWYXSNFSA-N | $3.0{\times}10^1$ | | HSDB (2015) | Q | 99 |
| 2,2,4-trimethyl-1,3-pentanediol diisobutyrate $C_{16}H_{30}O_4$ [6846-50-0] OMVSWZDEEGIJJI-UHFFFAOYSA-N | $9.0{\times}10^{-1}$ | | HSDB (2015) | Q | 99 |
| nonanedioic acid, dibutyl ester $C_{17}H_{32}O_4$ (dibutyl azelate) [2917-73-9] RISLXYINQFKFRL-UHFFFAOYSA-N | $8.2{\times}10^{-1}$ | | HSDB (2015) | Q | 99 |
| isopropyl myristate $C_{17}H_{34}O_2$ [110-27-0] AXISYYRBXTVTFY-UHFFFAOYSA-N | $4.2{\times}10^{-4}$ | | HSDB (2015) | Q | 99 |
| decanedioic acid, dibutyl ester $C_{18}H_{34}O_4$ [109-43-3] PYGXAGIECVVIOZ-UHFFFAOYSA-N | $2.0{\times}10^2$ $2.1{\times}10^2$ $1.8{\times}10^1$ | | Duchowicz et al. (2020) HSDB (2015) Duchowicz et al. (2020) | V V Q | 186 |
| diethylene glycol dibenzoate $C_{18}H_{18}O_5$ [120-55-8] NXQMCAOPTPLPRL-UHFFFAOYSA-N | $3.3{\times}10^6$ | | HSDB (2015) | Q | 99 |
| 12-hydroxy-9-octadecenoic acid, methyl ester $C_{19}H_{36}O_3$ (ricinoleic acid, methyl ester) [141-24-2] XKGDWZQXVZSXAO-RAXLEYEMSA-N | $6.7{\times}10^1$ | | HSDB (2015) | Q | 99 |
| chrysanthemumic acid 2,4-dimethylbenzyl ester $C_{19}H_{26}O_2$ (dimethrin) [70-38-2] FHNKBSDJERHDHZ-UHFFFAOYSA-N | $1.3{\times}10^{-1}$ | | HSDB (2015) | Q | 99 |



Table A3.8: Esters (RCOOR) (…continued)

| Substance<br>Formula<br>(Trivial Name)<br>[CAS Registry Number]<br>InChIKey | $H_s^{cp}$<br>(at $T^{\ominus}$)<br>$\left[\dfrac{\mathrm{mol}}{\mathrm{m^3\,Pa}}\right]$ | $\dfrac{\mathrm{d}\ln H_s^{cp}}{\mathrm{d}(1/T)}$<br><br>[K] | Reference | Type | Note |
|---|---|---|---|---|---|
| tributyl acetylcitrate<br>$C_{20}H_{34}O_8$<br>(acetyl tributyl citrate)<br>[77-90-7]<br>QZCLKYGREBVARF-UHFFFAOYSA-N | $2.6\times10^4$ | | HSDB (2015) | Q | 99 |
| hexanedioic acid,<br>bis[2-(2-butoxyethoxy)ethyl] ester<br>$C_{22}H_{42}O_8$<br>(bis(2-(2-butoxyethoxy)ethyl)<br>adipate)<br>[141-17-3]<br>SCABKEBYDRTODC-UHFFFAOYSA-N | $3.2\times10^7$ | | HSDB (2015) | Q | 99 |
| 1,2-benzenedicarboxylic acid,<br>decyl octyl ester<br>$C_{26}H_{42}O_4$<br>[119-07-3]<br>LVAGMBHLXLZJKZ-UHFFFAOYSA-N | $4.7\times10^{-1}$ | | HSDB (2015) | Q | 99 |
| phthalic acid, isodecyl octyl ester<br>$C_{26}H_{42}O_4$<br>[1330-96-7]<br>UXLBXTIBYMXRNX-UHFFFAOYSA-N | $4.7\times10^{-1}$ | | HSDB (2015) | Q | 99 |
| diisononyl hexahydrophthalate<br>$C_{26}H_{48}O_4$<br>[166412-78-8]<br>HORIEOQXBKUKGQ-UHFFFAOYSA-N | $1.4\times10^{-1}$ | | HSDB (2015) | Q | 99 |
| decanedioic acid, bis(2-ethylhexyl)<br>ester<br>$C_{26}H_{50}O_4$<br>(bis(2-ethylhexyl) sebacate)<br>[122-62-3]<br>VJHINFRRDQUWOJ-UHFFFAOYSA-N | $1.2\times10^{-1}$ | | HSDB (2015) | Q | 99 |
| glycerol tricaprylate<br>$C_{27}H_{50}O_6$<br>(tricaprylin)<br>[538-23-8]<br>VLPFTAMPNXLGLX-UHFFFAOYSA-N | $3.9\times10^2$ | | HSDB (2015) | Q | 99 |
| tris(2-ethylhexyl) trimellitate<br>$C_{33}H_{54}O_6$<br>[3319-31-1]<br>KRADHMIOFJQKEZ-UHFFFAOYSA-N | $2.2\times10^1$ | | HSDB (2015) | Q | 99 |
| MCM:CHOOCH2OOH<br>$C_2H_4O_4$<br>RHCLSHTZJFAXOU-UHFFFAOYSA-N | $3.8\times10^3$<br>$7.4\times10^2$<br>$6.3\times10^1$ | | Wang et al. (2017)<br>Wang et al. (2017)<br>Wang et al. (2017) | Q<br>Q<br>Q | 80, 238<br>80, 239<br>80, 240 |



Table A3.8: Esters (RCOOR) (...continued)

| Substance Formula (Trivial Name) [CAS Registry Number] InChIKey | $H_s^{cp}$ (at $T^\ominus$) $\left[\dfrac{\mathrm{mol}}{\mathrm{m^3\,Pa}}\right]$ | $\dfrac{\mathrm{d}\ln H_s^{cp}}{\mathrm{d}(1/T)}$ [K] | Reference | Type | Note |
|---|---|---|---|---|---|
| MCM:CHOOCHO | 3.3 | | Wang et al. (2017) | Q | 80, 238 |
| $C_2H_2O_3$ | $2.5\times10^1$ | | Wang et al. (2017) | Q | 80, 239 |
| VGGRCVDNFAQIKO-UHFFFAOYSA-N | $1.1\times10^{-1}$ | | Wang et al. (2017) | Q | 80, 240 |
| MCM:CHOOMCO3H | $4.6\times10^4$ | | Wang et al. (2017) | Q | 80, 238 |
| $C_3H_4O_5$ | $2.0\times10^3$ | | Wang et al. (2017) | Q | 80, 239 |
| NYMJKVCDMUPZFI-UHFFFAOYSA-N | $3.8\times10^1$ | | Wang et al. (2017) | Q | 80, 240 |
| MCM:ETHFORMOOH | $3.6\times10^3$ | | Wang et al. (2017) | Q | 80, 238 |
| $C_3H_6O_4$ | $2.9\times10^2$ | | Wang et al. (2017) | Q | 80, 239 |
| DJECRZORXDSOEF-UHFFFAOYSA-N | $2.5\times10^1$ | | Wang et al. (2017) | Q | 80, 240 |
| MCM:MECOFOROOH | $1.6\times10^5$ | | Wang et al. (2017) | Q | 80, 238 |
| $C_3H_4O_5$ | $3.6\times10^4$ | | Wang et al. (2017) | Q | 80, 239 |
| TUWVUCWHUYJQAO-UHFFFAOYSA-N | $2.0\times10^2$ | | Wang et al. (2017) | Q | 80, 240 |
| MCM:METACETHO | 2.2 | | Wang et al. (2017) | Q | 80, 238 |
| $C_3H_4O_3$ | $4.4\times10^1$ | | Wang et al. (2017) | Q | 80, 239 |
| ORWKVZNEPHTCQE-UHFFFAOYSA-N | $1.4\times10^{-1}$ | | Wang et al. (2017) | Q | 80, 240 |
| MCM:METACETO2H | $2.6\times10^3$ | | Wang et al. (2017) | Q | 80, 238 |
| $C_3H_6O_4$ | $1.4\times10^3$ | | Wang et al. (2017) | Q | 80, 239 |
| AIBRVUZTVKELQP-UHFFFAOYSA-N | $4.5\times10^1$ | | Wang et al. (2017) | Q | 80, 240 |
| MCM:MMCF | $1.7\times10^1$ | | Wang et al. (2017) | Q | 80, 238 |
| $C_3H_4O_4$ | $7.8\times10^1$ | | Wang et al. (2017) | Q | 80, 239 |
| IIHOUWJOSQHHHH-UHFFFAOYSA-N | $3.0\times10^{-1}$ | | Wang et al. (2017) | Q | 80, 240 |
| MCM:MMCFOOH | $1.2\times10^6$ | | Wang et al. (2017) | Q | 80, 238 |
| $C_3H_4O_6$ | $5.9\times10^5$ | | Wang et al. (2017) | Q | 80, 239 |
| LQGPNTLGGGGYKM-UHFFFAOYSA-N | $4.1\times10^2$ | | Wang et al. (2017) | Q | 80, 240 |
| MCM:MOCOCH2OOH | $2.6\times10^3$ | | Wang et al. (2017) | Q | 80, 238 |
| $C_3H_6O_4$ | $2.5\times10^2$ | | Wang et al. (2017) | Q | 80, 239 |
| OQIIXPHPHGKJRP-UHFFFAOYSA-N | $5.3\times10^1$ | | Wang et al. (2017) | Q | 80, 240 |
| MCM:ACEC2H4OOH | $2.1\times10^3$ | | Wang et al. (2017) | Q | 80, 238 |
| $C_4H_8O_4$ | $2.0\times10^3$ | | Wang et al. (2017) | Q | 80, 239 |
| JNHNRXKAKFYQGM-UHFFFAOYSA-N | $1.9\times10^1$ | | Wang et al. (2017) | Q | 80, 240 |
| MCM:ACETMECO3H | $3.2\times10^4$ | | Wang et al. (2017) | Q | 80, 238 |
| $C_4H_6O_5$ | $3.5\times10^3$ | | Wang et al. (2017) | Q | 80, 239 |
| UCKKJLQTXHILBL-UHFFFAOYSA-N | $7.8\times10^1$ | | Wang et al. (2017) | Q | 80, 240 |
| MCM:COO2C3CO3H | $3.2\times10^4$ | | Wang et al. (2017) | Q | 80, 238 |
| $C_4H_6O_5$ | $2.9\times10^3$ | | Wang et al. (2017) | Q | 80, 239 |
| CKNWOSHDVKEZAO-UHFFFAOYSA-N | $2.0\times10^1$ | | Wang et al. (2017) | Q | 80, 240 |
| MCM:COO2C4OOH | $2.1\times10^3$ | | Wang et al. (2017) | Q | 80, 238 |
| $C_4H_8O_4$ | $1.6\times10^3$ | | Wang et al. (2017) | Q | 80, 239 |
| KATJWBCCJXQUDM-UHFFFAOYSA-N | $4.4\times10^2$ | | Wang et al. (2017) | Q | 80, 240 |



Table A3.8: Esters (RCOOR) (...continued)

| Substance Formula (Trivial Name) [CAS Registry Number] InChIKey | $H_s^{cp}$ (at $T^\ominus$) $\left[\dfrac{\text{mol}}{\text{m}^3\,\text{Pa}}\right]$ | $\dfrac{\text{d}\ln H_s^{cp}}{\text{d}(1/T)}$ [K] | Reference | Type | Note |
|---|---|---|---|---|---|
| MCM:EOCOCH2OOH | $2.1\times10^3$ | | Wang et al. (2017) | Q | 80, 238 |
| $C_4H_8O_4$ | $1.1\times10^2$ | | Wang et al. (2017) | Q | 80, 239 |
| ZDXGLRBKCRTGQP-UHFFFAOYSA-N | $3.3\times10^1$ | | Wang et al. (2017) | Q | 80, 240 |
| MCM:ETHACETOOH | $2.4\times10^3$ | | Wang et al. (2017) | Q | 80, 238 |
| $C_4H_8O_4$ | $2.7\times10^2$ | | Wang et al. (2017) | Q | 80, 239 |
| FITNUWWXWGTMFO-UHFFFAOYSA-N | $2.8\times10^1$ | | Wang et al. (2017) | Q | 80, 240 |
| MCM:IPRFORMOOH | $2.0\times10^3$ | | Wang et al. (2017) | Q | 80, 238 |
| $C_4H_8O_4$ | $5.1\times10^1$ | | Wang et al. (2017) | Q | 80, 239 |
| DHNDYBGDIQCRQB-UHFFFAOYSA-N | 1.7 | | Wang et al. (2017) | Q | 80, 240 |
| maleic anhydride | $1.9\times10^1$ | | Wang et al. (2017) | Q | 80, 238 |
| $C_4H_2O_3$ | $1.2\times10^5$ | | Wang et al. (2017) | Q | 80, 239 |
| (MCM:MALANHY) | 9.6 | | Wang et al. (2017) | Q | 80, 240 |
| [108-31-6] | | | | | |
| FPYJFEHAWHCUMM-UHFFFAOYSA-N | | | | | |
| MCM:MECOACEOOH | $1.1\times10^5$ | | Wang et al. (2017) | Q | 80, 238 |
| $C_4H_6O_5$ | $6.6\times10^4$ | | Wang et al. (2017) | Q | 80, 239 |
| ZCPJTWSQAFRVMG-UHFFFAOYSA-N | $5.3\times10^2$ | | Wang et al. (2017) | Q | 80, 240 |
| MCM:ACCOETOOH | $8.7\times10^4$ | | Wang et al. (2017) | Q | 80, 238 |
| $C_5H_8O_5$ | $5.8\times10^5$ | | Wang et al. (2017) | Q | 80, 239 |
| MRPVYADEHOUTJI-UHFFFAOYSA-N | $8.1\times10^3$ | | Wang et al. (2017) | Q | 80, 240 |
| MCM:ACCOMECO3H | $1.3\times10^6$ | | Wang et al. (2017) | Q | 80, 238 |
| $C_5H_6O_6$ | $8.5\times10^5$ | | Wang et al. (2017) | Q | 80, 239 |
| OSZJUWKUAMKWAM-UHFFFAOYSA-N | $1.3\times10^3$ | | Wang et al. (2017) | Q | 80, 240 |
| MCM:ACETC2CO3H | $2.5\times10^4$ | | Wang et al. (2017) | Q | 80, 238 |
| $C_5H_8O_5$ | $3.2\times10^3$ | | Wang et al. (2017) | Q | 80, 239 |
| LEMDJBHCHPODAJ-UHFFFAOYSA-N | $5.0\times10^1$ | | Wang et al. (2017) | Q | 80, 240 |
| MCM:ACETCOC2H5 | 1.2 | | Wang et al. (2017) | Q | 80, 238 |
| $C_5H_8O_3$ | $2.5\times10^1$ | | Wang et al. (2017) | Q | 80, 239 |
| KLUDQUOLAFVLOL-UHFFFAOYSA-N | 2.4 | | Wang et al. (2017) | Q | 80, 240 |
| MCM:BUFORMOOH | $2.6\times10^3$ | | Wang et al. (2017) | Q | 80, 238 |
| $C_5H_{10}O_4$ | $9.8\times10^1$ | | Wang et al. (2017) | Q | 80, 239 |
| HTFWPRQCJNWCFL-UHFFFAOYSA-N | $1.3\times10^1$ | | Wang et al. (2017) | Q | 80, 240 |
| MCM:COO2C4CO3H | $2.5\times10^4$ | | Wang et al. (2017) | Q | 80, 238 |
| $C_5H_8O_5$ | $2.5\times10^3$ | | Wang et al. (2017) | Q | 80, 239 |
| XFYUESMQFXHHIL-UHFFFAOYSA-N | $1.6\times10^2$ | | Wang et al. (2017) | Q | 80, 240 |
| MCM:IPRACBCO3H | $2.8\times10^4$ | | Wang et al. (2017) | Q | 80, 238 |
| $C_5H_8O_5$ | $6.9\times10^2$ | | Wang et al. (2017) | Q | 80, 239 |
| IEXBKLGSFAHQBW-UHFFFAOYSA-N | 6.6 | | Wang et al. (2017) | Q | 80, 240 |
| MCM:IPRACBOOH | $1.9\times10^3$ | | Wang et al. (2017) | Q | 80, 238 |
| $C_5H_{10}O_4$ | $7.3\times10^2$ | | Wang et al. (2017) | Q | 80, 239 |
| HJBGHGSQOFAQDB-UHFFFAOYSA-N | 5.3 | | Wang et al. (2017) | Q | 80, 240 |



Table A3.8: Esters (RCOOR) (...continued)

| Substance Formula (Trivial Name) [CAS Registry Number] InChIKey | $H_s^{cp}$ (at $T^\ominus$) $\left[\dfrac{\mathrm{mol}}{\mathrm{m^3\,Pa}}\right]$ | $\dfrac{\mathrm{d\ln}H_s^{cp}}{\mathrm{d}(1/T)}$ [K] | Reference | Type | Note |
|---|---|---|---|---|---|
| MCM:IPRACOOH $C_5H_{10}O_4$ UMMAXRSQAGVERV-UHFFFAOYSA-N | $1.3\times10^3$ $5.9\times10^1$ 1.8 | | Wang et al. (2017) Wang et al. (2017) Wang et al. (2017) | Q Q Q | 80, 238 80, 239 80, 240 |
| MCM:MMALANHY $C_5H_4O_3$ AYKYXWQEBUNJCN-UHFFFAOYSA-N | $1.3\times10^1$ $2.1\times10^5$ 8.7 | | Wang et al. (2017) Wang et al. (2017) Wang et al. (2017) | Q Q Q | 80, 238 80, 239 80, 240 |
| MCM:MTBEAALOOH $C_5H_{10}O_4$ OJPAWJIXAIXDOD-UHFFFAOYSA-N | $1.6\times10^3$ $2.3\times10^2$ $5.1\times10^1$ | | Wang et al. (2017) Wang et al. (2017) Wang et al. (2017) | Q Q Q | 80, 238 80, 239 80, 240 |
| MCM:MTBEALCO3H $C_5H_8O_5$ GVHAIIVWIJOUAR-UHFFFAOYSA-N | $2.3\times10^4$ $1.1\times10^2$ $2.8\times10^1$ | | Wang et al. (2017) Wang et al. (2017) Wang et al. (2017) | Q Q Q | 80, 238 80, 239 80, 240 |
| MCM:NPRACAOOH $C_5H_{10}O_4$ BAISPQOWBPLMQI-UHFFFAOYSA-N | $1.9\times10^3$ $7.4\times10^2$ $2.3\times10^1$ | | Wang et al. (2017) Wang et al. (2017) Wang et al. (2017) | Q Q Q | 80, 238 80, 239 80, 240 |
| MCM:NPRACBOOH $C_5H_{10}O_4$ KUTULJFZWKOEBJ-UHFFFAOYSA-N | $1.9\times10^3$ $1.5\times10^2$ 9.8 | | Wang et al. (2017) Wang et al. (2017) Wang et al. (2017) | Q Q Q | 80, 238 80, 239 80, 240 |
| MCM:NPRACCOOH $C_5H_{10}O_4$ GYDFEBDSBUDYEN-UHFFFAOYSA-N | $1.7\times10^3$ $1.0\times10^3$ $5.4\times10^2$ | | Wang et al. (2017) Wang et al. (2017) Wang et al. (2017) | Q Q Q | 80, 238 80, 239 80, 240 |
| MCM:PRCOFORM $C_5H_8O_3$ JKMPLJUAEDQBOK-UHFFFAOYSA-N | 1.6 $1.4\times10^1$ $1.4\times10^{-1}$ | | Wang et al. (2017) Wang et al. (2017) Wang et al. (2017) | Q Q Q | 80, 238 80, 239 80, 240 |
| MCM:PRCOFOROOH $C_5H_8O_5$ VVRRHCGIFBEYON-UHFFFAOYSA-N | $1.2\times10^5$ $2.4\times10^5$ $9.1\times10^2$ | | Wang et al. (2017) Wang et al. (2017) Wang et al. (2017) | Q Q Q | 80, 238 80, 239 80, 240 |
| MCM:PRCOOMOOH $C_5H_{10}O_4$ JALZRJAYIBRIKJ-UHFFFAOYSA-N | $1.7\times10^3$ $4.9\times10^2$ $2.2\times10^1$ | | Wang et al. (2017) Wang et al. (2017) Wang et al. (2017) | Q Q Q | 80, 238 80, 239 80, 240 |
| MCM:TLFUONE $C_5H_6O_2$ BGLUXFNVVSVEET-UHFFFAOYSA-N | $3.2\times10^{-1}$ $1.4\times10^2$ $8.7\times10^2$ | | Wang et al. (2017) Wang et al. (2017) Wang et al. (2017) | Q Q Q | 80, 238 80, 239 80, 240 |
| MCM:ACCOPROOH $C_6H_{10}O_5$ HNMHVARGKGGINS-UHFFFAOYSA-N | $8.1\times10^4$ $2.1\times10^5$ $1.5\times10^4$ | | Wang et al. (2017) Wang et al. (2017) Wang et al. (2017) | Q Q Q | 80, 238 80, 239 80, 240 |
| MCM:ACETCOC3H7 $C_6H_{10}O_3$ BVQHHUQLZPXYAQ-UHFFFAOYSA-N | 1.0 $1.3\times10^1$ 2.4 | | Wang et al. (2017) Wang et al. (2017) Wang et al. (2017) | Q Q Q | 80, 238 80, 239 80, 240 |





Table A3.8: Esters (RCOOR) (...continued)

| Substance Formula (Trivial Name) [CAS Registry Number] InChIKey | $H_s^{cp}$ (at $T^\ominus$) $\left[\dfrac{\text{mol}}{\text{m}^3\,\text{Pa}}\right]$ | $\dfrac{\text{d}\ln H_s^{cp}}{\text{d}(1/T)}$ [K] | Reference | Type | Note |
|---|---|---|---|---|---|
| MCM:BOCOCH2OOH | $1.1\times10^3$ | | Wang et al. (2017) | Q | 80, 238 |
| $C_6H_{12}O_4$ | $3.5\times10^1$ | | Wang et al. (2017) | Q | 80, 239 |
| DQCMTBDKSIDVDF-UHFFFAOYSA-N | $1.7\times10^1$ | | Wang et al. (2017) | Q | 80, 240 |
| MCM:DMMALANHY | 8.1 | | Wang et al. (2017) | Q | 80, 238 |
| $C_6H_6O_3$ | $1.7\times10^5$ | | Wang et al. (2017) | Q | 80, 239 |
| MFGALGYVFGDXIX-UHFFFAOYSA-N | 8.7 | | Wang et al. (2017) | Q | 80, 240 |
| MCM:EBFUONE | $3.0\times10^{-1}$ | | Wang et al. (2017) | Q | 80, 238 |
| $C_6H_8O_2$ | $7.8\times10^1$ | | Wang et al. (2017) | Q | 80, 239 |
| GOUILHYTHSOMQJ-UHFFFAOYSA-N | $4.9\times10^2$ | | Wang et al. (2017) | Q | 80, 240 |
| MCM:EMALANHY | $1.0\times10^1$ | | Wang et al. (2017) | Q | 80, 238 |
| $C_6H_6O_3$ | $1.1\times10^5$ | | Wang et al. (2017) | Q | 80, 239 |
| AXGOOCLYBPQWNG-UHFFFAOYSA-N | 5.8 | | Wang et al. (2017) | Q | 80, 240 |
| MCM:MCOOTBOOH | $1.1\times10^3$ | | Wang et al. (2017) | Q | 80, 238 |
| $C_6H_{12}O_4$ | $2.8\times10^2$ | | Wang et al. (2017) | Q | 80, 239 |
| GZZAVFPVGHHZEL-UHFFFAOYSA-N | $3.6\times10^1$ | | Wang et al. (2017) | Q | 80, 240 |
| MCM:MXYFUONE | $2.2\times10^{-1}$ | | Wang et al. (2017) | Q | 80, 238 |
| $C_6H_8O_2$ | $9.1\times10^1$ | | Wang et al. (2017) | Q | 80, 239 |
| SAXRUMLUKZBSTO-UHFFFAOYSA-N | $3.3\times10^2$ | | Wang et al. (2017) | Q | 80, 240 |
| MCM:NBUACAOOH | $1.7\times10^3$ | | Wang et al. (2017) | Q | 80, 238 |
| $C_6H_{12}O_4$ | $5.9\times10^2$ | | Wang et al. (2017) | Q | 80, 239 |
| VUJYCIVMLKMZJB-UHFFFAOYSA-N | $3.6\times10^2$ | | Wang et al. (2017) | Q | 80, 240 |
| MCM:NBUACBOOH | $1.7\times10^3$ | | Wang et al. (2017) | Q | 80, 238 |
| $C_6H_{12}O_4$ | $4.4\times10^2$ | | Wang et al. (2017) | Q | 80, 239 |
| ZDNJENJHBNLWSZ-UHFFFAOYSA-N | 6.9 | | Wang et al. (2017) | Q | 80, 240 |
| MCM:NBUACCOOH | $1.7\times10^3$ | | Wang et al. (2017) | Q | 80, 238 |
| $C_6H_{12}O_4$ | $1.1\times10^2$ | | Wang et al. (2017) | Q | 80, 239 |
| CCXFYGGLIMLVCI-UHFFFAOYSA-N | $1.1\times10^1$ | | Wang et al. (2017) | Q | 80, 240 |
| MCM:PRCOOMCO3H | $2.0\times10^4$ | | Wang et al. (2017) | Q | 80, 238 |
| $C_6H_{10}O_5$ | $8.7\times10^2$ | | Wang et al. (2017) | Q | 80, 239 |
| NIESGQMUWIIUQP-UHFFFAOYSA-N | 2.1 | | Wang et al. (2017) | Q | 80, 240 |
| MCM:SBUACAOOH | $1.1\times10^3$ | | Wang et al. (2017) | Q | 80, 238 |
| $C_6H_{12}O_4$ | $3.6\times10^1$ | | Wang et al. (2017) | Q | 80, 239 |
| XEFYSNUZBIEBQZ-UHFFFAOYSA-N | 1.0 | | Wang et al. (2017) | Q | 80, 240 |
| MCM:SBUACBOOH | $1.8\times10^3$ | | Wang et al. (2017) | Q | 80, 238 |
| $C_6H_{12}O_4$ | $3.6\times10^2$ | | Wang et al. (2017) | Q | 80, 239 |
| SHPWQWOSZRJPCY-UHFFFAOYSA-N | $1.3\times10^1$ | | Wang et al. (2017) | Q | 80, 240 |
| MCM:TBUACCO3H | $1.6\times10^4$ | | Wang et al. (2017) | Q | 80, 238 |
| $C_6H_{10}O_5$ | $1.2\times10^2$ | | Wang et al. (2017) | Q | 80, 239 |
| CTLNPRHGCLAPPZ-UHFFFAOYSA-N | 2.7 | | Wang et al. (2017) | Q | 80, 240 |





Table A3.8: Esters (RCOOR) (. . . continued)

| Substance<br>Formula<br>(Trivial Name)<br>[CAS Registry Number]<br>InChIKey | $H_s^{cp}$<br>(at $T^\ominus$)<br>$\left[\dfrac{\mathrm{mol}}{\mathrm{m^3\,Pa}}\right]$ | $\dfrac{\mathrm{d\ln}H_s^{cp}}{\mathrm{d}(1/T)}$<br><br>[K] | Reference | Type | Note |
|---|---|---|---|---|---|
| MCM:TMB1FUONE<br>$C_6H_8O_2$<br>GLCOTRLUPUIAFI-UHFFFAOYSA-N | $2.2\times10^{-1}$<br>$1.3\times10^2$<br>$3.9\times10^3$ | | Wang et al. (2017)<br>Wang et al. (2017)<br>Wang et al. (2017) | Q<br>Q<br>Q | 80, 238<br>80, 239<br>80, 240 |
| MCM:IPBFUONE<br>$C_7H_{10}O_2$<br>LWFKYUIZWLTJCA-UHFFFAOYSA-N | $2.8\times10^{-1}$<br>$6.9\times10^1$<br>$4.6\times10^2$ | | Wang et al. (2017)<br>Wang et al. (2017)<br>Wang et al. (2017) | Q<br>Q<br>Q | 80, 238<br>80, 239<br>80, 240 |
| MCM:IPMALANHY<br>$C_7H_8O_3$<br>QSWLSAYLEATCSH-UHFFFAOYSA-N | 9.3<br>$6.3\times10^4$<br>2.8 | | Wang et al. (2017)<br>Wang et al. (2017)<br>Wang et al. (2017) | Q<br>Q<br>Q | 80, 238<br>80, 239<br>80, 240 |
| MCM:MEBFUONE<br>$C_7H_{10}O_2$<br>UVERQUDHQJYTIT-UHFFFAOYSA-N | $2.0\times10^{-1}$<br>$5.4\times10^1$<br>$2.0\times10^2$ | | Wang et al. (2017)<br>Wang et al. (2017)<br>Wang et al. (2017) | Q<br>Q<br>Q | 80, 238<br>80, 239<br>80, 240 |
| MCM:PBFUONE<br>$C_7H_{10}O_2$<br>CLCLZCBNFSGUOD-UHFFFAOYSA-N | $2.3\times10^{-1}$<br>$5.5\times10^1$<br>$4.2\times10^2$ | | Wang et al. (2017)<br>Wang et al. (2017)<br>Wang et al. (2017) | Q<br>Q<br>Q | 80, 238<br>80, 239<br>80, 240 |
| MCM:PMALANHY<br>$C_7H_8O_3$<br>LPFJFXRQANKTRA-UHFFFAOYSA-N | 8.7<br>$6.0\times10^4$<br>4.2 | | Wang et al. (2017)<br>Wang et al. (2017)<br>Wang et al. (2017) | Q<br>Q<br>Q | 80, 238<br>80, 239<br>80, 240 |
| MCM:TMB2FUONE<br>$C_7H_{10}O_2$<br>OSFZDFIHIYXIEL-UHFFFAOYSA-N | $1.5\times10^{-1}$<br>$9.1\times10^1$<br>$1.4\times10^3$ | | Wang et al. (2017)<br>Wang et al. (2017)<br>Wang et al. (2017) | Q<br>Q<br>Q | 80, 238<br>80, 239<br>80, 240 |
| MCM:C1013OOH<br>$C_{10}H_{18}O_4$<br>BQYOGPDLQGPEQM-UHFFFAOYSA-N | $2.1\times10^3$<br>$9.8\times10^2$<br>$2.9\times10^2$ | | Wang et al. (2017)<br>Wang et al. (2017)<br>Wang et al. (2017) | Q<br>Q<br>Q | 80, 238<br>80, 239<br>80, 240 |
| MCM:C1013CO3H<br>$C_{11}H_{18}O_5$<br>ZWRGNVXICSEPNH-UHFFFAOYSA-N | $2.2\times10^4$<br>$6.3\times10^2$<br>$1.3\times10^2$ | | Wang et al. (2017)<br>Wang et al. (2017)<br>Wang et al. (2017) | Q<br>Q<br>Q | 80, 238<br>80, 239<br>80, 240 |
| MCM:PXYFUONE<br>$C_5H_6O_2$<br>VGHBEMPMIVEGJP-UHFFFAOYSA-N | $2.3\times10^{-1}$<br>$3.0\times10^2$<br>$4.5\times10^2$ | | Wang et al. (2017)<br>Wang et al. (2017)<br>Wang et al. (2017) | Q<br>Q<br>Q | 80, 238<br>80, 239<br>80, 240 |
| MCM:OXYFUONE<br>$C_6H_8O_2$<br>QHQDWCHELGHSTO-UHFFFAOYSA-N | $1.6\times10^{-1}$<br>$2.8\times10^2$<br>$1.5\times10^3$ | | Wang et al. (2017)<br>Wang et al. (2017)<br>Wang et al. (2017) | Q<br>Q<br>Q | 80, 238<br>80, 239<br>80, 240 |
| MCM:CHOOCH2OH<br>$C_2H_4O_3$<br>APUQIHKCZFWODD-UHFFFAOYSA-N | $1.4\times10^2$<br>$6.5\times10^1$<br>$1.7\times10^1$ | | Wang et al. (2017)<br>Wang et al. (2017)<br>Wang et al. (2017) | Q<br>Q<br>Q | 80, 238<br>80, 239<br>80, 240 |
| MCM:ETHFORMOH<br>$C_3H_6O_3$<br>JRUKHAIVAGVYRP-UHFFFAOYSA-N | $1.4\times10^2$<br>$6.2\times10^1$<br>7.3 | | Wang et al. (2017)<br>Wang et al. (2017)<br>Wang et al. (2017) | Q<br>Q<br>Q | 80, 238<br>80, 239<br>80, 240 |





Table A3.8: Esters (RCOOR) (...continued)

| Substance Formula (Trivial Name) [CAS Registry Number] InChIKey | $H_s^{cp}$ (at $T^{\ominus}$) $\left[\dfrac{\mathrm{mol}}{\mathrm{m^3\,Pa}}\right]$ | $\dfrac{\mathrm{d}\ln H_s^{cp}}{\mathrm{d}(1/T)}$ [K] | Reference | Type | Note |
|---|---|---|---|---|---|
| MCM:ETOHOCHO | $1.6\times10^2$ | | Wang et al. (2017) | Q | 80, 238 |
| $C_3H_6O_3$ | $3.3\times10^2$ | | Wang et al. (2017) | Q | 80, 239 |
| UKQJDWBNQNAJHB-UHFFFAOYSA-N | $2.2\times10^2$ | | Wang et al. (2017) | Q | 80, 240 |
| MCM:METACETOH | $9.8\times10^1$ | | Wang et al. (2017) | Q | 80, 238 |
| $C_3H_6O_3$ | $1.1\times10^2$ | | Wang et al. (2017) | Q | 80, 239 |
| JYVNDCLJHKQUHE-UHFFFAOYSA-N | $1.0\times10^1$ | | Wang et al. (2017) | Q | 80, 240 |
| MCM:MMCFOH | $4.6\times10^4$ | | Wang et al. (2017) | Q | 80, 238 |
| $C_3H_4O_5$ | $1.4\times10^5$ | | Wang et al. (2017) | Q | 80, 239 |
| XDYLYGBFMHJJOW-UHFFFAOYSA-N | $1.6\times10^2$ | | Wang et al. (2017) | Q | 80, 240 |
| MCM:MOXCOCH2OH | $5.0$ | | Wang et al. (2017) | Q | 80, 238 |
| $C_3H_6O_3$ | $1.5\times10^2$ | | Wang et al. (2017) | Q | 80, 239 |
| GSJFXBNYJCXDGI-UHFFFAOYSA-N | $2.0\times10^1$ | | Wang et al. (2017) | Q | 80, 240 |
| MCM:ACEETOHOOH | $8.3\times10^6$ | | Wang et al. (2017) | Q | 80, 238 |
| $C_4H_8O_5$ | $2.9\times10^4$ | | Wang et al. (2017) | Q | 80, 239 |
| QQYYYRFGUJQKRH-UHFFFAOYSA-N | $2.3\times10^4$ | | Wang et al. (2017) | Q | 80, 240 |
| MCM:BZFUOH | $4.8\times10^6$ | | Wang et al. (2017) | Q | 80, 238 |
| $C_4H_6O_4$ | $3.3\times10^7$ | | Wang et al. (2017) | Q | 80, 239 |
| SGMJBNSHAZVGMC-UHFFFAOYSA-N | $1.7\times10^7$ | | Wang et al. (2017) | Q | 80, 240 |
| MCM:BZFUOOH | $7.8\times10^9$ | | Wang et al. (2017) | Q | 80, 238 |
| $C_4H_6O_5$ | $8.9\times10^7$ | | Wang et al. (2017) | Q | 80, 239 |
| LZJWYIKGTDPLAQ-UHFFFAOYSA-N | $3.2\times10^6$ | | Wang et al. (2017) | Q | 80, 240 |
| MCM:EOX2COMEOH | $4.0$ | | Wang et al. (2017) | Q | 80, 238 |
| $C_4H_8O_3$ | $7.1\times10^1$ | | Wang et al. (2017) | Q | 80, 239 |
| ZANNOFHADGWOLI-UHFFFAOYSA-N | $1.2\times10^1$ | | Wang et al. (2017) | Q | 80, 240 |
| MCM:ETACETOH | $9.6\times10^1$ | | Wang et al. (2017) | Q | 80, 238 |
| $C_4H_8O_3$ | $3.6\times10^2$ | | Wang et al. (2017) | Q | 80, 239 |
| HXDLWJWIAHWIKI-UHFFFAOYSA-N | $3.4\times10^2$ | | Wang et al. (2017) | Q | 80, 240 |
| MCM:ETHACETOH | $8.9\times10^1$ | | Wang et al. (2017) | Q | 80, 238 |
| $C_4H_8O_3$ | $7.1\times10^1$ | | Wang et al. (2017) | Q | 80, 239 |
| MLAFRLBDNVSLPE-UHFFFAOYSA-N | $7.4$ | | Wang et al. (2017) | Q | 80, 240 |
| MCM:HOACETEOOH | $3.0\times10^5$ | | Wang et al. (2017) | Q | 80, 238 |
| $C_4H_8O_5$ | $5.6\times10^5$ | | Wang et al. (2017) | Q | 80, 239 |
| QWYGOSGRTMPWCI-UHFFFAOYSA-N | $5.9\times10^3$ | | Wang et al. (2017) | Q | 80, 240 |
| MCM:IPRFORMOH | $7.4\times10^1$ | | Wang et al. (2017) | Q | 80, 238 |
| $C_4H_8O_3$ | $3.2\times10^1$ | | Wang et al. (2017) | Q | 80, 239 |
| SVJMGLNDGCVDIT-UHFFFAOYSA-N | $2.9$ | | Wang et al. (2017) | Q | 80, 240 |
| MCM:MALANHY2OH | $7.1\times10^8$ | | Wang et al. (2017) | Q | 80, 238 |
| $C_4H_4O_5$ | $4.8\times10^8$ | | Wang et al. (2017) | Q | 80, 239 |
| BOGVTNYNTGOONP-UHFFFAOYSA-N | $3.9\times10^6$ | | Wang et al. (2017) | Q | 80, 240 |





Table A3.8: Esters (RCOOR) (...continued)

| Substance Formula (Trivial Name) [CAS Registry Number] InChIKey | $H_s^{cp}$ (at $T^\ominus$) $\left[\dfrac{\text{mol}}{\text{m}^3\,\text{Pa}}\right]$ | $\dfrac{\text{d}\ln H_s^{cp}}{\text{d}(1/T)}$ [K] | Reference | Type | Note |
|---|---|---|---|---|---|
| MCM:MALANHYOOH $C_4H_4O_6$ UPSCPEYLZZUIND-UHFFFAOYSA-N | $1.2\times10^{12}$ $1.0\times10^{9}$ $1.3\times10^{7}$ | | Wang et al. (2017) Wang et al. (2017) Wang et al. (2017) | Q Q Q | 80, 238 80, 239 80, 240 |
| MCM:PROL2FORM $C_4H_8O_3$ CRMUFGZQBODVGS-UHFFFAOYSA-N | $1.5\times10^{2}$ $2.6\times10^{2}$ $1.3\times10^{2}$ | | Wang et al. (2017) Wang et al. (2017) Wang et al. (2017) | Q Q Q | 80, 238 80, 239 80, 240 |
| MCM:BUOHFORM $C_5H_{10}O_3$ KIZCCPGSHHAFRX-UHFFFAOYSA-N | $8.7\times10^{1}$ $3.0\times10^{1}$ 4.6 | | Wang et al. (2017) Wang et al. (2017) Wang et al. (2017) | Q Q Q | 80, 238 80, 239 80, 240 |
| MCM:IPRACBOH $C_5H_{10}O_3$ YJNKLTDJZSXVHQ-UHFFFAOYSA-N | $1.0\times10^{2}$ $2.0\times10^{2}$ $2.7\times10^{2}$ | | Wang et al. (2017) Wang et al. (2017) Wang et al. (2017) | Q Q Q | 80, 238 80, 239 80, 240 |
| MCM:IPRACOH $C_5H_{10}O_3$ MYOAZFOMGTTYOD-UHFFFAOYSA-N | $5.0\times10^{1}$ $4.1\times10^{1}$ 1.2 | | Wang et al. (2017) Wang et al. (2017) Wang et al. (2017) | Q Q Q | 80, 238 80, 239 80, 240 |
| MCM:MMALNHY2OH $C_5H_6O_5$ DGWWNMWYNUWPBC-UHFFFAOYSA-N | $3.9\times10^{8}$ $6.8\times10^{8}$ $1.4\times10^{6}$ | | Wang et al. (2017) Wang et al. (2017) Wang et al. (2017) | Q Q Q | 80, 238 80, 239 80, 240 |
| MCM:MMALNHYOOH $C_5H_6O_6$ JNCHKPBVXWMPAJ-UHFFFAOYSA-N | $6.3\times10^{11}$ $6.8\times10^{8}$ $7.3\times10^{5}$ | | Wang et al. (2017) Wang et al. (2017) Wang et al. (2017) | Q Q Q | 80, 238 80, 239 80, 240 |
| MCM:MTBEACHOHO $C_5H_{10}O_3$ GQMJADBSHHFADW-UHFFFAOYSA-N | $8.1\times10^{1}$ $7.6\times10^{1}$ 8.3 | | Wang et al. (2017) Wang et al. (2017) Wang et al. (2017) | Q Q Q | 80, 238 80, 239 80, 240 |
| MCM:NPRACAOH $C_5H_{10}O_3$ PPPFYBPQAPISCT-UHFFFAOYSA-N | $1.0\times10^{2}$ $3.2\times10^{2}$ $2.3\times10^{2}$ | | Wang et al. (2017) Wang et al. (2017) Wang et al. (2017) | Q Q Q | 80, 238 80, 239 80, 240 |
| MCM:NPRACBOH $C_5H_{10}O_3$ VEGXEWGKYMMJKP-UHFFFAOYSA-N | $7.3\times10^{1}$ $4.7\times10^{1}$ 3.2 | | Wang et al. (2017) Wang et al. (2017) Wang et al. (2017) | Q Q Q | 80, 238 80, 239 80, 240 |
| MCM:NPRACCOH $C_5H_{10}O_3$ DOUBAFNWVFAWEC-UHFFFAOYSA-N | $6.0\times10^{1}$ $6.3\times10^{2}$ $9.8\times10^{1}$ | | Wang et al. (2017) Wang et al. (2017) Wang et al. (2017) | Q Q Q | 80, 238 80, 239 80, 240 |
| MCM:PXYFUOH $C_5H_8O_4$ OHTGZAWPVDWARE-UHFFFAOYSA-N | $2.7\times10^{6}$ $4.3\times10^{7}$ $3.0\times10^{6}$ | | Wang et al. (2017) Wang et al. (2017) Wang et al. (2017) | Q Q Q | 80, 238 80, 239 80, 240 |
| MCM:PXYFUOOH $C_5H_8O_5$ IEFDVDGOIVKGRZ-UHFFFAOYSA-N | $4.5\times10^{9}$ $4.9\times10^{7}$ $1.7\times10^{6}$ | | Wang et al. (2017) Wang et al. (2017) Wang et al. (2017) | Q Q Q | 80, 238 80, 239 80, 240 |





Table A3.8: Esters (RCOOR) (. . . continued)

| Substance Formula (Trivial Name) [CAS Registry Number] InChIKey | $H_s^{cp}$ (at $T^{\ominus}$) $\left[\dfrac{\text{mol}}{\text{m}^3\,\text{Pa}}\right]$ | $\dfrac{\text{d}\ln H_s^{cp}}{\text{d}(1/T)}$ [K] | Reference | Type | Note |
|---|---|---|---|---|---|
| MCM:TLFUOH | $4.5\times10^6$ | | Wang et al. (2017) | Q | 80, 238 |
| $C_5H_8O_4$ | $3.7\times10^7$ | | Wang et al. (2017) | Q | 80, 239 |
| JYHWQRJRDKSSIF-UHFFFAOYSA-N | $7.4\times10^6$ | | Wang et al. (2017) | Q | 80, 240 |
| MCM:TLFUOOH | $7.3\times10^9$ | | Wang et al. (2017) | Q | 80, 238 |
| $C_5H_8O_5$ | $1.0\times10^8$ | | Wang et al. (2017) | Q | 80, 239 |
| GNXADMJANDYCAE-UHFFFAOYSA-N | $2.9\times10^6$ | | Wang et al. (2017) | Q | 80, 240 |
| MCM:BOX2COMOH | 2.9 | | Wang et al. (2017) | Q | 80, 238 |
| $C_6H_{12}O_3$ | $2.9\times10^1$ | | Wang et al. (2017) | Q | 80, 239 |
| VFGRALUHHHDIQI-UHFFFAOYSA-N | 9.8 | | Wang et al. (2017) | Q | 80, 240 |
| MCM:DMMALYOH | $2.2\times10^8$ | | Wang et al. (2017) | Q | 80, 238 |
| $C_6H_8O_5$ | $1.2\times10^9$ | | Wang et al. (2017) | Q | 80, 239 |
| KIOPGGAGDHZEGB-UHFFFAOYSA-N | $4.9\times10^4$ | | Wang et al. (2017) | Q | 80, 240 |
| MCM:DMMALYOOH | $3.5\times10^{11}$ | | Wang et al. (2017) | Q | 80, 238 |
| $C_6H_8O_6$ | $8.9\times10^8$ | | Wang et al. (2017) | Q | 80, 239 |
| VRKLPKITGMCVQP-UHFFFAOYSA-N | $1.4\times10^5$ | | Wang et al. (2017) | Q | 80, 240 |
| MCM:EBFUOH | $4.1\times10^6$ | | Wang et al. (2017) | Q | 80, 238 |
| $C_6H_{10}O_4$ | $3.5\times10^7$ | | Wang et al. (2017) | Q | 80, 239 |
| AQSJHOPJZAKOAP-UHFFFAOYSA-N | $3.7\times10^6$ | | Wang et al. (2017) | Q | 80, 240 |
| MCM:EBFUOOH | $6.0\times10^9$ | | Wang et al. (2017) | Q | 80, 238 |
| $C_6H_{10}O_5$ | $8.5\times10^7$ | | Wang et al. (2017) | Q | 80, 239 |
| KIZOVCBYCQGADC-UHFFFAOYSA-N | $7.6\times10^5$ | | Wang et al. (2017) | Q | 80, 240 |
| MCM:EMALNHY2OH | $3.6\times10^8$ | | Wang et al. (2017) | Q | 80, 238 |
| $C_6H_8O_5$ | $8.3\times10^8$ | | Wang et al. (2017) | Q | 80, 239 |
| GPJQUVLKFIAXFZ-UHFFFAOYSA-N | $1.7\times10^6$ | | Wang et al. (2017) | Q | 80, 240 |
| MCM:EMALNHYOOH | $5.0\times10^{11}$ | | Wang et al. (2017) | Q | 80, 238 |
| $C_6H_8O_6$ | $8.5\times10^8$ | | Wang et al. (2017) | Q | 80, 239 |
| LSSHMOINJMVLHQ-UHFFFAOYSA-N | $6.3\times10^5$ | | Wang et al. (2017) | Q | 80, 240 |
| MCM:MXYFUOH | $2.5\times10^6$ | | Wang et al. (2017) | Q | 80, 238 |
| $C_6H_{10}O_4$ | $3.7\times10^7$ | | Wang et al. (2017) | Q | 80, 239 |
| WCMYKJKBZGIGND-UHFFFAOYSA-N | $3.0\times10^6$ | | Wang et al. (2017) | Q | 80, 240 |
| MCM:MXYFUOOH | $4.2\times10^9$ | | Wang et al. (2017) | Q | 80, 238 |
| $C_6H_{10}O_5$ | $2.8\times10^7$ | | Wang et al. (2017) | Q | 80, 239 |
| WCIMCWKIICZLLM-UHFFFAOYSA-N | $3.6\times10^6$ | | Wang et al. (2017) | Q | 80, 240 |
| MCM:NBUACAOH | $5.6\times10^1$ | | Wang et al. (2017) | Q | 80, 238 |
| $C_6H_{12}O_3$ | $6.5\times10^2$ | | Wang et al. (2017) | Q | 80, 239 |
| KLUHZXMBIDAHSJ-UHFFFAOYSA-N | $6.3\times10^1$ | | Wang et al. (2017) | Q | 80, 240 |
| MCM:NBUACBOH | $8.0\times10^1$ | | Wang et al. (2017) | Q | 80, 238 |
| $C_6H_{12}O_3$ | $2.1\times10^2$ | | Wang et al. (2017) | Q | 80, 239 |
| BZLQSYFOTWOIDC-UHFFFAOYSA-N | $5.1\times10^1$ | | Wang et al. (2017) | Q | 80, 240 |





Table A3.8: Esters (RCOOR) (...continued)

| Substance Formula (Trivial Name) [CAS Registry Number] InChIKey | $H_s^{cp}$ (at $T^\ominus$) $\left[\dfrac{\mathrm{mol}}{\mathrm{m}^3\,\mathrm{Pa}}\right]$ | $\dfrac{\mathrm{d}\ln H_s^{cp}}{\mathrm{d}(1/T)}$ [K] | Reference | Type | Note |
|---|---|---|---|---|---|
| MCM:NBUACCOH $C_6H_{12}O_3$ BMNRJWUOBYYCRX-UHFFFAOYSA-N | $5.6\times10^1$ $3.6\times10^1$ $2.7$ | | Wang et al. (2017) Wang et al. (2017) Wang et al. (2017) | Q Q Q | 80, 238 80, 239 80, 240 |
| MCM:OXYFUOH $C_6H_{10}O_4$ GZKXVWYDQQSODI-UHFFFAOYSA-N | $1.5\times10^6$ $3.2\times10^7$ $1.7\times10^4$ | | Wang et al. (2017) Wang et al. (2017) Wang et al. (2017) | Q Q Q | 80, 238 80, 239 80, 240 |
| MCM:OXYFUOOH $C_6H_{10}O_5$ DHHFHIZBMAEZJT-UHFFFAOYSA-N | $2.4\times10^9$ $2.3\times10^7$ $3.6\times10^4$ | | Wang et al. (2017) Wang et al. (2017) Wang et al. (2017) | Q Q Q | 80, 238 80, 239 80, 240 |
| MCM:PRCOOETOH $C_6H_{12}O_3$ GIOCILWWMFZESP-UHFFFAOYSA-N | $6.9\times10^1$ $1.3\times10^2$ $1.1\times10^2$ | | Wang et al. (2017) Wang et al. (2017) Wang et al. (2017) | Q Q Q | 80, 238 80, 239 80, 240 |
| MCM:SBUACAOH $C_6H_{12}O_3$ WAXXMBOGFVJIHX-UHFFFAOYSA-N | $4.0\times10^1$ $3.2\times10^1$ $7.6\times10^{-1}$ | | Wang et al. (2017) Wang et al. (2017) Wang et al. (2017) | Q Q Q | 80, 238 80, 239 80, 240 |
| MCM:SBUACBOH $C_6H_{12}O_3$ BCWWODMTUXMSAB-UHFFFAOYSA-N | $9.6\times10^1$ $1.7\times10^2$ $1.1\times10^2$ | | Wang et al. (2017) Wang et al. (2017) Wang et al. (2017) | Q Q Q | 80, 238 80, 239 80, 240 |
| MCM:TBOCOCH2OH $C_6H_{12}O_3$ WINGEFIITRDOLJ-UHFFFAOYSA-N | $2.0$ $1.7\times10^1$ $1.7$ | | Wang et al. (2017) Wang et al. (2017) Wang et al. (2017) | Q Q Q | 80, 238 80, 239 80, 240 |
| MCM:TBUACOH $C_6H_{12}O_3$ GWOKHTMCBUZSOP-UHFFFAOYSA-N | $4.9\times10^1$ $9.1\times10^1$ $1.5\times10^1$ | | Wang et al. (2017) Wang et al. (2017) Wang et al. (2017) | Q Q Q | 80, 238 80, 239 80, 240 |
| MCM:TMB1FUOH $C_6H_{10}O_4$ AUTABLFJFWRKGZ-UHFFFAOYSA-N | $2.5\times10^6$ $2.2\times10^7$ $1.7\times10^6$ | | Wang et al. (2017) Wang et al. (2017) Wang et al. (2017) | Q Q Q | 80, 238 80, 239 80, 240 |
| MCM:TMB1FUOOH $C_6H_{10}O_5$ NFODFJMABSYJFC-UHFFFAOYSA-N | $1.8\times10^8$ $7.4\times10^7$ $4.2\times10^6$ | | Wang et al. (2017) Wang et al. (2017) Wang et al. (2017) | Q Q Q | 80, 238 80, 239 80, 240 |
| MCM:BOXCOOLOOH $C_7H_{14}O_5$ PDOIGHDIMNJPNL-UHFFFAOYSA-N | $2.1\times10^5$ $2.4\times10^5$ $1.0\times10^2$ | | Wang et al. (2017) Wang et al. (2017) Wang et al. (2017) | Q Q Q | 80, 238 80, 239 80, 240 |
| MCM:IPBFUOH $C_7H_{12}O_4$ ZIPHTXAVHSYMQR-UHFFFAOYSA-N | $3.9\times10^6$ $3.0\times10^7$ $5.3\times10^6$ | | Wang et al. (2017) Wang et al. (2017) Wang et al. (2017) | Q Q Q | 80, 238 80, 239 80, 240 |
| MCM:IPBFUOOH $C_7H_{12}O_5$ HVQWCSPFZPPNOE-UHFFFAOYSA-N | $5.4\times10^9$ $7.8\times10^7$ $2.1\times10^6$ | | Wang et al. (2017) Wang et al. (2017) Wang et al. (2017) | Q Q Q | 80, 238 80, 239 80, 240 |



Table A3.8: Esters (RCOOR) (...continued)

| Substance<br>Formula<br>(Trivial Name)<br>[CAS Registry Number]<br>InChIKey | $H_s^{cp}$<br>(at $T^{\ominus}$)<br>$\left[\dfrac{\mathrm{mol}}{\mathrm{m^3\,Pa}}\right]$ | $\dfrac{\mathrm{d}\ln H_s^{cp}}{\mathrm{d}(1/T)}$<br><br>[K] | Reference | Type | Note |
|---|---|---|---|---|---|
| MCM:IPMLNHY2OH | $3.2\times10^8$ | | Wang et al. (2017) | Q | 80, 238 |
| $C_7H_{10}O_5$ | $1.2\times10^9$ | | Wang et al. (2017) | Q | 80, 239 |
| RBIJYTIBBLMIRU-UHFFFAOYSA-N | $4.3\times10^5$ | | Wang et al. (2017) | Q | 80, 240 |
| MCM:IPMLNHYOOH | $4.7\times10^{11}$ | | Wang et al. (2017) | Q | 80, 238 |
| $C_7H_{10}O_6$ | $1.1\times10^9$ | | Wang et al. (2017) | Q | 80, 239 |
| LGGLTRMSNOIPLU-UHFFFAOYSA-N | $9.3\times10^5$ | | Wang et al. (2017) | Q | 80, 240 |
| MCM:MEBFUOH | $2.2\times10^6$ | | Wang et al. (2017) | Q | 80, 238 |
| $C_7H_{12}O_4$ | $2.2\times10^7$ | | Wang et al. (2017) | Q | 80, 239 |
| SYHJYGJIPQZFIO-UHFFFAOYSA-N | $1.6\times10^6$ | | Wang et al. (2017) | Q | 80, 240 |
| MCM:MEBFUOOH | $3.2\times10^9$ | | Wang et al. (2017) | Q | 80, 238 |
| $C_7H_{12}O_5$ | $1.7\times10^7$ | | Wang et al. (2017) | Q | 80, 239 |
| VRHMFTJWRSNJLN-UHFFFAOYSA-N | $2.2\times10^6$ | | Wang et al. (2017) | Q | 80, 240 |
| MCM:PBFUOH | $3.2\times10^6$ | | Wang et al. (2017) | Q | 80, 238 |
| $C_7H_{12}O_4$ | $2.4\times10^7$ | | Wang et al. (2017) | Q | 80, 239 |
| DMMDFVMQHRHVSU-UHFFFAOYSA-N | $5.6\times10^6$ | | Wang et al. (2017) | Q | 80, 240 |
| MCM:PBFUOOH | $5.3\times10^9$ | | Wang et al. (2017) | Q | 80, 238 |
| $C_7H_{12}O_5$ | $6.2\times10^7$ | | Wang et al. (2017) | Q | 80, 239 |
| KAMUACRVJWJEFX-UHFFFAOYSA-N | $6.3\times10^5$ | | Wang et al. (2017) | Q | 80, 240 |
| MCM:PMALNHY2OH | $2.8\times10^8$ | | Wang et al. (2017) | Q | 80, 238 |
| $C_7H_{10}O_5$ | $8.9\times10^8$ | | Wang et al. (2017) | Q | 80, 239 |
| PVDNZCIFUUKKBS-UHFFFAOYSA-N | $1.3\times10^6$ | | Wang et al. (2017) | Q | 80, 240 |
| MCM:PMALNHYOOH | $4.1\times10^{11}$ | | Wang et al. (2017) | Q | 80, 238 |
| $C_7H_{10}O_6$ | $8.0\times10^8$ | | Wang et al. (2017) | Q | 80, 239 |
| VKSMRBCXNQEUFP-UHFFFAOYSA-N | $4.4\times10^5$ | | Wang et al. (2017) | Q | 80, 240 |
| MCM:PRCOOPROL | $6.5\times10^1$ | | Wang et al. (2017) | Q | 80, 238 |
| $C_7H_{14}O_3$ | $1.3\times10^2$ | | Wang et al. (2017) | Q | 80, 239 |
| QXXGFEMAULVMST-UHFFFAOYSA-N | $4.5\times10^1$ | | Wang et al. (2017) | Q | 80, 240 |
| MCM:TMB2FUOH | $1.4\times10^6$ | | Wang et al. (2017) | Q | 80, 238 |
| $C_7H_{12}O_4$ | $1.8\times10^7$ | | Wang et al. (2017) | Q | 80, 239 |
| YEQCQVSGZKCWAA-UHFFFAOYSA-N | $1.3\times10^6$ | | Wang et al. (2017) | Q | 80, 240 |
| MCM:TMB2FUOOH | $2.2\times10^9$ | | Wang et al. (2017) | Q | 80, 238 |
| $C_7H_{12}O_5$ | $1.2\times10^7$ | | Wang et al. (2017) | Q | 80, 239 |
| MINPHQZFUPUEBD-UHFFFAOYSA-N | $3.6\times10^6$ | | Wang et al. (2017) | Q | 80, 240 |
| MCM:C1013OH | $7.1\times10^1$ | | Wang et al. (2017) | Q | 80, 238 |
| $C_{10}H_{18}O_3$ | $5.4\times10^2$ | | Wang et al. (2017) | Q | 80, 239 |
| OYGVMQZTRRRABW-UHFFFAOYSA-N | $3.0\times10^1$ | | Wang et al. (2017) | Q | 80, 240 |
| MCM:ALCOMOXOOH | $2.3\times10^6$ | | Wang et al. (2017) | Q | 80, 238 |
| $C_3H_4O_5$ | $1.6\times10^5$ | | Wang et al. (2017) | Q | 80, 239 |
| DRQRCDXSDDWNEU-UHFFFAOYSA-N | $9.3\times10^2$ | | Wang et al. (2017) | Q | 80, 240 |





Table A3.8: Esters (RCOOR) (. . . continued)

| Substance Formula (Trivial Name) [CAS Registry Number] InChIKey | $H_s^{cp}$ (at $T^\ominus$) $\left[\dfrac{\mathrm{mol}}{\mathrm{m^3\,Pa}}\right]$ | $\dfrac{\mathrm{d}\ln H_s^{cp}}{\mathrm{d}(1/T)}$ [K] | Reference | Type | Note |
|---|---|---|---|---|---|
| MCM:DFC $C_3H_2O_5$ DOMHJQJKOPZSOM-UHFFFAOYSA-N | $1.0\times10^3$ $6.9\times10^3$ $1.2\times10^{-1}$ | | Wang et al. (2017) Wang et al. (2017) Wang et al. (2017) | Q Q Q | 80, 238 80, 239 80, 240 |
| MCM:MOXY2CHO $C_3H_4O_3$ MFRMAQFRVTUENW-UHFFFAOYSA-N | $3.7\times10^1$ $1.1\times10^2$ $1.1\times10^1$ | | Wang et al. (2017) Wang et al. (2017) Wang et al. (2017) | Q Q Q | 80, 238 80, 239 80, 240 |
| MCM:MOXYCOCHO $C_3H_4O_3$ KFKXSMSQHIOMSO-UHFFFAOYSA-N | $3.1\times10^1$ $3.2\times10^1$ $6.5\times10^{-1}$ | | Wang et al. (2017) Wang et al. (2017) Wang et al. (2017) | Q Q Q | 80, 238 80, 239 80, 240 |
| MCM:ACETETCHO $C_4H_6O_3$ GUPGZURVZDIQPM-UHFFFAOYSA-N | $2.5\times10^1$ $1.8\times10^2$ $1.7\times10^1$ | | Wang et al. (2017) Wang et al. (2017) Wang et al. (2017) | Q Q Q | 80, 238 80, 239 80, 240 |
| MCM:CO14O3CHO $C_4H_4O_4$ ZFWWLKBCHQUFEJ-UHFFFAOYSA-N | $2.3\times10^4$ $1.2\times10^4$ $8.0\times10^1$ | | Wang et al. (2017) Wang et al. (2017) Wang et al. (2017) | Q Q Q | 80, 238 80, 239 80, 240 |
| MCM:COO2C3CHO $C_4H_6O_3$ AGADEVQOWQDDFX-UHFFFAOYSA-N | $2.5\times10^1$ $5.4\times10^1$ $1.1\times10^1$ | | Wang et al. (2017) Wang et al. (2017) Wang et al. (2017) | Q Q Q | 80, 238 80, 239 80, 240 |
| MCM:EOCOCHO $C_4H_6O_3$ DBPFRRFGLYGEJI-UHFFFAOYSA-N | $2.5\times10^1$ $1.7\times10^1$ $8.7\times10^{-1}$ | | Wang et al. (2017) Wang et al. (2017) Wang et al. (2017) | Q Q Q | 80, 238 80, 239 80, 240 |
| MCM:ACCOMECHO $C_5H_6O_4$ GRUDDVNYUQQBCU-UHFFFAOYSA-N | $1.1\times10^3$ $1.6\times10^4$ $9.8\times10^1$ | | Wang et al. (2017) Wang et al. (2017) Wang et al. (2017) | Q Q Q | 80, 238 80, 239 80, 240 |
| MCM:ACEC2CHO $C_5H_8O_3$ PRSPLAWXBFRHKV-UHFFFAOYSA-N | $2.0\times10^1$ $1.7\times10^2$ $7.4\times10^1$ | | Wang et al. (2017) Wang et al. (2017) Wang et al. (2017) | Q Q Q | 80, 238 80, 239 80, 240 |
| MCM:COO2C4CHO $C_5H_8O_3$ DLZVZNAPRCRXEG-UHFFFAOYSA-N | $2.0\times10^1$ $1.3\times10^2$ $3.5\times10^1$ | | Wang et al. (2017) Wang et al. (2017) Wang et al. (2017) | Q Q Q | 80, 238 80, 239 80, 240 |
| MCM:IPRACBCHO $C_5H_8O_3$ FXPPNKAYSGWCQG-UHFFFAOYSA-N | $2.3\times10^1$ $3.9\times10^1$ $5.0$ | | Wang et al. (2017) Wang et al. (2017) Wang et al. (2017) | Q Q Q | 80, 238 80, 239 80, 240 |
| MCM:MTBEACHO13 $C_5H_8O_3$ MDWWHOIDIKOPHH-UHFFFAOYSA-N | $1.9\times10^1$ $5.8$ $1.7$ | | Wang et al. (2017) Wang et al. (2017) Wang et al. (2017) | Q Q Q | 80, 238 80, 239 80, 240 |
| MCM:BOXCOALOOH $C_6H_{10}O_5$ MZUJTBRWESAMQA-UHFFFAOYSA-N | $1.4\times10^6$ $2.0\times10^4$ $4.4\times10^1$ | | Wang et al. (2017) Wang et al. (2017) Wang et al. (2017) | Q Q Q | 80, 238 80, 239 80, 240 |



Table A3.8: Esters (RCOOR) (...continued)

| Substance<br>Formula<br>(Trivial Name)<br>[CAS Registry Number]<br>InChIKey | $H_s^{cp}$<br>(at $T^{\ominus}$)<br>$\left[\dfrac{\text{mol}}{\text{m}^3\,\text{Pa}}\right]$ | $\dfrac{\text{d}\ln H_s^{cp}}{\text{d}(1/T)}$<br><br>[K] | Reference | Type | Note |
|---|---|---|---|---|---|
| MCM:BOXCOCHO | $1.8\times10^1$ | | Wang et al. (2017) | Q | 80, 238 |
| $C_6H_{10}O_3$ | 6.0 | | Wang et al. (2017) | Q | 80, 239 |
| NRYDRJHYTRBBEA-UHFFFAOYSA-N | $1.5\times10^{-1}$ | | Wang et al. (2017) | Q | 80, 240 |
| MCM:C14O3ECHO | $1.7\times10^4$ | | Wang et al. (2017) | Q | 80, 238 |
| $C_6H_8O_4$ | $3.3\times10^3$ | | Wang et al. (2017) | Q | 80, 239 |
| FVLOJCOPOCRBFF-UHFFFAOYSA-N | 5.6 | | Wang et al. (2017) | Q | 80, 240 |
| MCM:PRCOOMCHO | $1.8\times10^1$ | | Wang et al. (2017) | Q | 80, 238 |
| $C_6H_{10}O_3$ | $4.7\times10^1$ | | Wang et al. (2017) | Q | 80, 239 |
| YEBNSRBQIIWQQC-UHFFFAOYSA-N | $1.0\times10^1$ | | Wang et al. (2017) | Q | 80, 240 |
| MCM:TBOCOCHO | $1.3\times10^1$ | | Wang et al. (2017) | Q | 80, 238 |
| $C_6H_{10}O_3$ | 3.6 | | Wang et al. (2017) | Q | 80, 239 |
| WDPZTIFGRQKSEN-UHFFFAOYSA-N | $1.4\times10^{-1}$ | | Wang et al. (2017) | Q | 80, 240 |
| MCM:TBUACCO | $1.3\times10^1$ | | Wang et al. (2017) | Q | 80, 238 |
| $C_6H_{10}O_3$ | 6.6 | | Wang et al. (2017) | Q | 80, 239 |
| BYYPPVMBDPUHHR-UHFFFAOYSA-N | 5.5 | | Wang et al. (2017) | Q | 80, 240 |
| MCM:C14O3IPCHO | $1.6\times10^4$ | | Wang et al. (2017) | Q | 80, 238 |
| $C_7H_{10}O_4$ | $2.3\times10^3$ | | Wang et al. (2017) | Q | 80, 239 |
| NPPWDFRKAMDHMM-UHFFFAOYSA-N | 7.8 | | Wang et al. (2017) | Q | 80, 240 |
| MCM:C14O3PCHO | $1.5\times10^4$ | | Wang et al. (2017) | Q | 80, 238 |
| $C_7H_{10}O_4$ | $2.1\times10^3$ | | Wang et al. (2017) | Q | 80, 239 |
| GNODSIBUYMXKGR-UHFFFAOYSA-N | 3.6 | | Wang et al. (2017) | Q | 80, 240 |
| MCM:C1014OOH | $5.3\times10^5$ | | Wang et al. (2017) | Q | 80, 238 |
| $C_{10}H_{18}O_5$ | $8.3\times10^4$ | | Wang et al. (2017) | Q | 80, 239 |
| REYZAYCUYYNPLJ-UHFFFAOYSA-N | $2.0\times10^3$ | | Wang et al. (2017) | Q | 80, 240 |
| MCM:BCALBOC | $2.1\times10^1$ | | Wang et al. (2017) | Q | 80, 238 |
| $C_{15}H_{24}O_3$ | $3.0\times10^1$ | | Wang et al. (2017) | Q | 80, 239 |
| FEPYZLRQFOQWRK-UHFFFAOYSA-N | $8.5\times10^2$ | | Wang et al. (2017) | Q | 80, 240 |
| MCM:CHOOMCO2H | $4.7\times10^5$ | | Wang et al. (2017) | Q | 80, 238 |
| $C_3H_6O_4$ | $1.9\times10^5$ | | Wang et al. (2017) | Q | 80, 239 |
| CBNAABWGSNQFTA-UHFFFAOYSA-N | $8.3\times10^4$ | | Wang et al. (2017) | Q | 80, 240 |
| MCM:C1014OH | $1.9\times10^4$ | | Wang et al. (2017) | Q | 80, 238 |
| $C_{10}H_{18}O_4$ | $2.1\times10^5$ | | Wang et al. (2017) | Q | 80, 239 |
| IJPJFXOOJHUBTK-UHFFFAOYSA-N | $9.1\times10^3$ | | Wang et al. (2017) | Q | 80, 240 |
| MCM:C152OH | $2.3\times10^7$ | | Wang et al. (2017) | Q | 80, 238 |
| $C_{15}H_{26}O_5$ | $1.0\times10^8$ | | Wang et al. (2017) | Q | 80, 239 |
| XZXFZFZFZSKVDT-UHFFFAOYSA-N | $1.7\times10^6$ | | Wang et al. (2017) | Q | 80, 240 |
| MCM:C152OOH | $2.0\times10^9$ | | Wang et al. (2017) | Q | 80, 238 |
| $C_{15}H_{26}O_6$ | $2.2\times10^8$ | | Wang et al. (2017) | Q | 80, 239 |
| PKQOWOPWICZSPN-UHFFFAOYSA-N | $5.1\times10^7$ | | Wang et al. (2017) | Q | 80, 240 |



Table A3.8: Esters (RCOOR) (...continued)

| Substance<br>Formula<br>(Trivial Name)<br>[CAS Registry Number]<br>InChIKey | $H_s^{cp}$<br>(at $T^\ominus$)<br>$\left[\dfrac{\mathrm{mol}}{\mathrm{m}^3\,\mathrm{Pa}}\right]$ | $\dfrac{\mathrm{d}\ln H_s^{cp}}{\mathrm{d}(1/T)}$<br><br>[K] | Reference | Type | Note |
|---|---|---|---|---|---|
| MCM:C23O3CHO | $1.3\times10^3$ | | Wang et al. (2017) | Q | 80, 238 |
| $C_4H_4O_4$ | $7.3\times10^3$ | | Wang et al. (2017) | Q | 80, 239 |
| RULCSXTUAHNYHJ-UHFFFAOYSA-N | $6.2\times10^{-1}$ | | Wang et al. (2017) | Q | 80, 240 |
| MCM:MCOCOMOOOH | $1.6\times10^6$ | | Wang et al. (2017) | Q | 80, 238 |
| $C_4H_6O_5$ | $2.5\times10^5$ | | Wang et al. (2017) | Q | 80, 239 |
| BJTKGFWDIAZUGI-UHFFFAOYSA-N | $1.2\times10^3$ | | Wang et al. (2017) | Q | 80, 240 |
| MCM:MCOCOMOX | $2.1\times10^1$ | | Wang et al. (2017) | Q | 80, 238 |
| $C_4H_6O_3$ | $3.0\times10^1$ | | Wang et al. (2017) | Q | 80, 239 |
| CWKLZLBVOJRSOM-UHFFFAOYSA-N | 4.3 | | Wang et al. (2017) | Q | 80, 240 |
| MCM:PRNFORMOOH | $2.2\times10^6$ | | Wang et al. (2017) | Q | 80, 238 |
| $C_4H_6O_5$ | $5.4\times10^3$ | | Wang et al. (2017) | Q | 80, 239 |
| LIFMXPSJWIBAGH-UHFFFAOYSA-N | $2.0\times10^2$ | | Wang et al. (2017) | Q | 80, 240 |
| MCM:PRONFORM | $2.5\times10^1$ | | Wang et al. (2017) | Q | 80, 238 |
| $C_4H_6O_3$ | $1.5\times10^2$ | | Wang et al. (2017) | Q | 80, 239 |
| WNVAJGCMEDTLIE-UHFFFAOYSA-N | $3.9\times10^1$ | | Wang et al. (2017) | Q | 80, 240 |
| MCM:ACCOCOMOOH | $6.8\times10^7$ | | Wang et al. (2017) | Q | 80, 238 |
| $C_5H_6O_6$ | $3.2\times10^6$ | | Wang et al. (2017) | Q | 80, 239 |
| WKJHUBLAYZIBAJ-UHFFFAOYSA-N | $1.3\times10^3$ | | Wang et al. (2017) | Q | 80, 240 |
| MCM:ACECOCOCH3 | $8.9\times10^2$ | | Wang et al. (2017) | Q | 80, 238 |
| $C_5H_6O_4$ | $9.1\times10^3$ | | Wang et al. (2017) | Q | 80, 239 |
| OBTVATRSWTWNGS-UHFFFAOYSA-N | $8.7\times10^{-1}$ | | Wang et al. (2017) | Q | 80, 240 |
| MCM:ACEPROPONE | $1.7\times10^1$ | | Wang et al. (2017) | Q | 80, 238 |
| $C_5H_8O_3$ | $1.8\times10^2$ | | Wang et al. (2017) | Q | 80, 239 |
| DBERHVIZRVGDFO-UHFFFAOYSA-N | $8.0\times10^1$ | | Wang et al. (2017) | Q | 80, 240 |
| MCM:ACPRONEOOH | $1.5\times10^6$ | | Wang et al. (2017) | Q | 80, 238 |
| $C_5H_8O_5$ | $5.5\times10^3$ | | Wang et al. (2017) | Q | 80, 239 |
| FOWIMBLOPIOWII-UHFFFAOYSA-N | $2.3\times10^2$ | | Wang et al. (2017) | Q | 80, 240 |
| MCM:C23O3CCO3H | $1.6\times10^7$ | | Wang et al. (2017) | Q | 80, 238 |
| $C_5H_6O_6$ | $5.5\times10^5$ | | Wang et al. (2017) | Q | 80, 239 |
| SPYAIQASSDQEIX-UHFFFAOYSA-N | $1.1\times10^3$ | | Wang et al. (2017) | Q | 80, 240 |
| MCM:C23O3MCOOH | $1.5\times10^6$ | | Wang et al. (2017) | Q | 80, 238 |
| $C_5H_8O_5$ | $6.6\times10^4$ | | Wang et al. (2017) | Q | 80, 239 |
| RVGUYYZMHBERKG-UHFFFAOYSA-N | $6.5\times10^2$ | | Wang et al. (2017) | Q | 80, 240 |
| MCM:PRNOCOMOOH | $1.3\times10^6$ | | Wang et al. (2017) | Q | 80, 238 |
| $C_5H_8O_5$ | $2.2\times10^5$ | | Wang et al. (2017) | Q | 80, 239 |
| GJTAOZDAMDVHTP-UHFFFAOYSA-N | $8.3\times10^3$ | | Wang et al. (2017) | Q | 80, 240 |
| MCM:ACBUOAOOH | $1.2\times10^6$ | | Wang et al. (2017) | Q | 80, 238 |
| $C_6H_{10}O_5$ | $1.1\times10^5$ | | Wang et al. (2017) | Q | 80, 239 |
| BDHJVYCZNHBETI-UHFFFAOYSA-N | $3.8\times10^3$ | | Wang et al. (2017) | Q | 80, 240 |





Table A3.8: Esters (RCOOR) (...continued)

| Substance<br>Formula<br>(Trivial Name)<br>[CAS Registry Number]<br>InChIKey | $H_s^{cp}$<br>(at $T^{\ominus}$)<br>$\left[\dfrac{\mathrm{mol}}{\mathrm{m^3\,Pa}}\right]$ | $\dfrac{\mathrm{d\ln}H_s^{cp}}{\mathrm{d}(1/T)}$<br><br>[K] | Reference | Type | Note |
|---|---|---|---|---|---|
| MCM:ACBUONBOOH | $1.2\times10^6$ | | Wang et al. (2017) | Q | 80, 238 |
| $C_6H_{10}O_5$ | $2.5\times10^3$ | | Wang et al. (2017) | Q | 80, 239 |
| RISONYRIGLPILV-UHFFFAOYSA-N | $1.1\times10^2$ | | Wang et al. (2017) | Q | 80, 240 |
| MCM:ACCOCOC2H5 | $6.9\times10^2$ | | Wang et al. (2017) | Q | 80, 238 |
| $C_6H_8O_4$ | $4.7\times10^3$ | | Wang et al. (2017) | Q | 80, 239 |
| WZGYRWNUCYXCFD-UHFFFAOYSA-N | $3.8\times10^{-1}$ | | Wang et al. (2017) | Q | 80, 240 |
| MCM:ACCOCOEOOH | $6.0\times10^7$ | | Wang et al. (2017) | Q | 80, 238 |
| $C_6H_8O_6$ | $1.5\times10^6$ | | Wang et al. (2017) | Q | 80, 239 |
| RKPSGJVLERUIIG-UHFFFAOYSA-N | $2.1\times10^2$ | | Wang et al. (2017) | Q | 80, 240 |
| MCM:ACCOPRONE | $6.9\times10^2$ | | Wang et al. (2017) | Q | 80, 238 |
| $C_6H_8O_4$ | $1.6\times10^4$ | | Wang et al. (2017) | Q | 80, 239 |
| RYLHMXPGTMYTTF-UHFFFAOYSA-N | $2.5\times10^2$ | | Wang et al. (2017) | Q | 80, 240 |
| MCM:ACEBUTBONE | $1.4\times10^1$ | | Wang et al. (2017) | Q | 80, 238 |
| $C_6H_{10}O_3$ | $8.1\times10^1$ | | Wang et al. (2017) | Q | 80, 239 |
| LHGWJCBYBIICPP-UHFFFAOYSA-N | $4.2\times10^1$ | | Wang et al. (2017) | Q | 80, 240 |
| MCM:ACEBUTONE | $1.4\times10^1$ | | Wang et al. (2017) | Q | 80, 238 |
| $C_6H_{10}O_3$ | $1.8\times10^2$ | | Wang et al. (2017) | Q | 80, 239 |
| NWCYECXHIYEBJE-UHFFFAOYSA-N | $5.4\times10^2$ | | Wang et al. (2017) | Q | 80, 240 |
| MCM:ACOMCOMOOH | $5.3\times10^7$ | | Wang et al. (2017) | Q | 80, 238 |
| $C_6H_8O_6$ | $5.4\times10^6$ | | Wang et al. (2017) | Q | 80, 239 |
| NKXSSYAJYGYQPN-UHFFFAOYSA-N | $4.7\times10^3$ | | Wang et al. (2017) | Q | 80, 240 |
| MCM:C23O3ECOOH | $1.2\times10^6$ | | Wang et al. (2017) | Q | 80, 238 |
| $C_6H_{10}O_5$ | $3.0\times10^4$ | | Wang et al. (2017) | Q | 80, 239 |
| NLYPTRYHUBHKKI-UHFFFAOYSA-N | $4.3\times10^2$ | | Wang et al. (2017) | Q | 80, 240 |
| MCM:C23O3EHO | $6.9\times10^2$ | | Wang et al. (2017) | Q | 80, 238 |
| $C_6H_8O_4$ | $4.6\times10^3$ | | Wang et al. (2017) | Q | 80, 239 |
| AIDIMUPPUNPUTI-UHFFFAOYSA-N | $3.7\times10^{-1}$ | | Wang et al. (2017) | Q | 80, 240 |
| MCM:C23O3MCO3H | $1.5\times10^7$ | | Wang et al. (2017) | Q | 80, 238 |
| $C_6H_8O_6$ | $1.1\times10^5$ | | Wang et al. (2017) | Q | 80, 239 |
| FIPISRZHVPVVNA-UHFFFAOYSA-N | $9.8\times10^2$ | | Wang et al. (2017) | Q | 80, 240 |
| MCM:C6O4KETOOH | $3.6\times10^{10}$ | | Wang et al. (2017) | Q | 80, 238 |
| $C_6H_6O_7$ | $3.8\times10^8$ | | Wang et al. (2017) | Q | 80, 239 |
| LLKUMVBDCSCXDR-UHFFFAOYSA-N | $1.1\times10^3$ | | Wang et al. (2017) | Q | 80, 240 |
| MCM:C6OTKETCO | $5.5\times10^5$ | | Wang et al. (2017) | Q | 80, 238 |
| $C_6H_6O_5$ | $1.0\times10^6$ | | Wang et al. (2017) | Q | 80, 239 |
| GAPVBMQBWAUBHC-UHFFFAOYSA-N | $3.2\times10^{-1}$ | | Wang et al. (2017) | Q | 80, 240 |
| MCM:C6OTKETOOH | $7.8\times10^8$ | | Wang et al. (2017) | Q | 80, 238 |
| $C_6H_8O_6$ | $8.3\times10^5$ | | Wang et al. (2017) | Q | 80, 239 |
| USMDUVUWRKQXQG-UHFFFAOYSA-N | $6.8\times10^3$ | | Wang et al. (2017) | Q | 80, 240 |



Table A3.8: Esters (RCOOR) (...continued)

| Substance Formula (Trivial Name) [CAS Registry Number] InChIKey | $H_s^{cp}$ (at $T^{\ominus}$) $\left[\dfrac{\mathrm{mol}}{\mathrm{m^3\,Pa}}\right]$ | $\dfrac{\mathrm{d}\ln H_s^{cp}}{\mathrm{d}(1/T)}$ [K] | Reference | Type | Note |
|---|---|---|---|---|---|
| MCM:C6OTRIKET | $1.0\times10^4$ | | Wang et al. (2017) | Q | 80, 238 |
| $C_6H_8O_4$ | $2.4\times10^4$ | | Wang et al. (2017) | Q | 80, 239 |
| XAHWCAXZWOTSBC-UHFFFAOYSA-N | $5.1\times10^2$ | | Wang et al. (2017) | Q | 80, 240 |
| MCM:CO356OCOOH | $6.0\times10^7$ | | Wang et al. (2017) | Q | 80, 238 |
| $C_6H_8O_6$ | $4.5\times10^6$ | | Wang et al. (2017) | Q | 80, 239 |
| MVKXXKVCXUKLOS-UHFFFAOYSA-N | $1.6\times10^2$ | | Wang et al. (2017) | Q | 80, 240 |
| MCM:SBUACCOOH | $8.1\times10^5$ | | Wang et al. (2017) | Q | 80, 238 |
| $C_6H_{10}O_5$ | $5.3\times10^2$ | | Wang et al. (2017) | Q | 80, 239 |
| DTGNYNGASXJUMP-UHFFFAOYSA-N | $1.3\times10^1$ | | Wang et al. (2017) | Q | 80, 240 |
| MCM:SBUACEONE | $1.6\times10^1$ | | Wang et al. (2017) | Q | 80, 238 |
| $C_6H_{10}O_3$ | $3.3\times10^1$ | | Wang et al. (2017) | Q | 80, 239 |
| ZKPTYCJWRHHBOW-UHFFFAOYSA-N | $2.0\times10^1$ | | Wang et al. (2017) | Q | 80, 240 |
| MCM:BOX2COMOOH | $9.1\times10^5$ | | Wang et al. (2017) | Q | 80, 238 |
| $C_7H_{12}O_5$ | $2.1\times10^4$ | | Wang et al. (2017) | Q | 80, 239 |
| YCDXONIQHOVSSF-UHFFFAOYSA-N | $3.2\times10^2$ | | Wang et al. (2017) | Q | 80, 240 |
| MCM:BOXCOCOME | $1.2\times10^1$ | | Wang et al. (2017) | Q | 80, 238 |
| $C_7H_{12}O_3$ | $6.3$ | | Wang et al. (2017) | Q | 80, 239 |
| ZAZUOXBHFXAWMD-UHFFFAOYSA-N | $1.3$ | | Wang et al. (2017) | Q | 80, 240 |
| MCM:C23O3ECO3H | $1.4\times10^7$ | | Wang et al. (2017) | Q | 80, 238 |
| $C_7H_{10}O_6$ | $5.1\times10^4$ | | Wang et al. (2017) | Q | 80, 239 |
| FWVKIPPVGIEQRB-UHFFFAOYSA-N | $8.1\times10^1$ | | Wang et al. (2017) | Q | 80, 240 |
| MCM:MC6OTKTOOH | $4.5\times10^8$ | | Wang et al. (2017) | Q | 80, 238 |
| $C_7H_{10}O_6$ | $9.3\times10^4$ | | Wang et al. (2017) | Q | 80, 239 |
| DJTBKMNUYULRIV-UHFFFAOYSA-N | $1.4\times10^3$ | | Wang et al. (2017) | Q | 80, 240 |
| MCM:MC6OTRIKET | $9.6\times10^3$ | | Wang et al. (2017) | Q | 80, 238 |
| $C_7H_{10}O_4$ | $5.9\times10^3$ | | Wang et al. (2017) | Q | 80, 239 |
| FJYCWZCSXDGURV-UHFFFAOYSA-N | $2.6\times10^2$ | | Wang et al. (2017) | Q | 80, 240 |
| MCM:PRCOOPRONE | $1.2\times10^1$ | | Wang et al. (2017) | Q | 80, 238 |
| $C_7H_{12}O_3$ | $5.0\times10^1$ | | Wang et al. (2017) | Q | 80, 239 |
| AIJLJYUDTAJRDN-UHFFFAOYSA-N | $3.8\times10^1$ | | Wang et al. (2017) | Q | 80, 240 |
| MCM:PRNOCOPOOH | $9.1\times10^5$ | | Wang et al. (2017) | Q | 80, 238 |
| $C_7H_{12}O_5$ | $6.8\times10^5$ | | Wang et al. (2017) | Q | 80, 239 |
| RGEURRORAZXRIV-UHFFFAOYSA-N | $1.7\times10^4$ | | Wang et al. (2017) | Q | 80, 240 |
| MCM:C1013CO | $1.6\times10^1$ | | Wang et al. (2017) | Q | 80, 238 |
| $C_{10}H_{16}O_3$ | $8.3\times10^1$ | | Wang et al. (2017) | Q | 80, 239 |
| DPRGOCIPALHGLN-UHFFFAOYSA-N | $6.5\times10^2$ | | Wang et al. (2017) | Q | 80, 240 |
| MCM:BZFUCO | $1.2\times10^4$ | | Wang et al. (2017) | Q | 80, 238 |
| $C_4H_4O_4$ | $4.2\times10^6$ | | Wang et al. (2017) | Q | 80, 239 |
| MFFNSLKCUZEAFI-UHFFFAOYSA-N | $1.1\times10^5$ | | Wang et al. (2017) | Q | 80, 240 |



Table A3.8: Esters (RCOOR) (...continued)

| Substance<br>Formula<br>(Trivial Name)<br>[CAS Registry Number]<br>InChIKey | $H_s^{cp}$<br>(at $T^{\ominus}$)<br>$\left[\dfrac{\text{mol}}{\text{m}^3\,\text{Pa}}\right]$ | $\dfrac{\text{d}\ln H_s^{cp}}{\text{d}(1/T)}$<br><br>[K] | Reference | Type | Note |
|---|---|---|---|---|---|
| MCM:MALNHYOHCO<br>$C_4H_2O_5$<br>DBWAPECJPVXLJZ-UHFFFAOYSA-N | $2.6\times10^7$<br>$1.0\times10^9$<br>$1.1\times10^5$ | | Wang et al. (2017)<br>Wang et al. (2017)<br>Wang et al. (2017) | Q<br>Q<br>Q | 80, 238<br>80, 239<br>80, 240 |
| MCM:ACPRONEOH<br>$C_5H_8O_4$<br>LBOLMEABDOYQGX-UHFFFAOYSA-N | $2.5\times10^3$<br>$1.3\times10^4$<br>$7.6\times10^1$ | | Wang et al. (2017)<br>Wang et al. (2017)<br>Wang et al. (2017) | Q<br>Q<br>Q | 80, 238<br>80, 239<br>80, 240 |
| MCM:C23O3MOH<br>$C_5H_8O_4$<br>WHFDSTOTVFOUIH-UHFFFAOYSA-N | $4.8\times10^4$<br>$2.0\times10^4$<br>$3.6\times10^2$ | | Wang et al. (2017)<br>Wang et al. (2017)<br>Wang et al. (2017) | Q<br>Q<br>Q | 80, 238<br>80, 239<br>80, 240 |
| MCM:ACBUONAOH<br>$C_6H_{10}O_4$<br>UGJVDYQVVBKWGL-UHFFFAOYSA-N | $4.5\times10^4$<br>$5.1\times10^4$<br>$2.4\times10^3$ | | Wang et al. (2017)<br>Wang et al. (2017)<br>Wang et al. (2017) | Q<br>Q<br>Q | 80, 238<br>80, 239<br>80, 240 |
| MCM:ACBUONBOH<br>$C_6H_{10}O_4$<br>GYQPXJWBTJDNGC-UHFFFAOYSA-N | $2.3\times10^3$<br>$6.8\times10^3$<br>$7.4\times10^1$ | | Wang et al. (2017)<br>Wang et al. (2017)<br>Wang et al. (2017) | Q<br>Q<br>Q | 80, 238<br>80, 239<br>80, 240 |
| MCM:C23O3EOH<br>$C_6H_{10}O_4$<br>BVDDSYZMRAYYIJ-UHFFFAOYSA-N | $4.5\times10^4$<br>$1.1\times10^4$<br>$2.3\times10^2$ | | Wang et al. (2017)<br>Wang et al. (2017)<br>Wang et al. (2017) | Q<br>Q<br>Q | 80, 238<br>80, 239<br>80, 240 |
| MCM:C6OTKETOH<br>$C_6H_8O_5$<br>NOUUUBJIFPBZMZ-UHFFFAOYSA-N | $4.4\times10^4$<br>$1.5\times10^6$<br>$2.0\times10^6$<br>$6.9\times10^2$ | 16000 | Wieser et al. (2023)<br>Wang et al. (2017)<br>Wang et al. (2017)<br>Wang et al. (2017) | Q<br>Q<br>Q<br>Q | 437<br>80, 238<br>80, 239<br>80, 240 |
| MCM:EBFUCO<br>$C_6H_8O_4$<br>QFGGPABGKQEYQI-UHFFFAOYSA-N | $1.0\times10^4$<br>$5.0\times10^6$<br>$7.3\times10^3$ | | Wang et al. (2017)<br>Wang et al. (2017)<br>Wang et al. (2017) | Q<br>Q<br>Q | 80, 238<br>80, 239<br>80, 240 |
| MCM:SBUACCOH<br>$C_6H_{10}O_4$<br>SUDJMSTXSVVFNQ-UHFFFAOYSA-N | $1.4\times10^3$<br>$4.5\times10^3$<br>$5.0\times10^1$ | | Wang et al. (2017)<br>Wang et al. (2017)<br>Wang et al. (2017) | Q<br>Q<br>Q | 80, 238<br>80, 239<br>80, 240 |
| MCM:IPBFUCO<br>$C_7H_{10}O_4$<br>NHSFIZKDULUUEP-UHFFFAOYSA-N | $9.6\times10^3$<br>$4.6\times10^6$<br>$6.9\times10^3$ | | Wang et al. (2017)<br>Wang et al. (2017)<br>Wang et al. (2017) | Q<br>Q<br>Q | 80, 238<br>80, 239<br>80, 240 |
| MCM:MC6OTKETOH<br>$C_7H_{10}O_5$<br>GQVIXHCRVNFYGI-UHFFFAOYSA-N | $8.3\times10^5$<br>$7.1\times10^5$<br>$1.0\times10^3$ | | Wang et al. (2017)<br>Wang et al. (2017)<br>Wang et al. (2017) | Q<br>Q<br>Q | 80, 238<br>80, 239<br>80, 240 |
| MCM:PBFUCO<br>$C_7H_{10}O_4$<br>PKKIVTPUHZVJRM-UHFFFAOYSA-N | $8.0\times10^3$<br>$3.6\times10^6$<br>$7.4\times10^3$ | | Wang et al. (2017)<br>Wang et al. (2017)<br>Wang et al. (2017) | Q<br>Q<br>Q | 80, 238<br>80, 239<br>80, 240 |
| MCM:C23O3CCHO<br>$C_5H_6O_4$<br>JQZIMULKJBYVEJ-UHFFFAOYSA-N | $1.5\times10^4$<br>$2.1\times10^4$<br>$1.5\times10^2$ | | Wang et al. (2017)<br>Wang et al. (2017)<br>Wang et al. (2017) | Q<br>Q<br>Q | 80, 238<br>80, 239<br>80, 240 |





Table A3.8: Esters (RCOOR) (...continued)

| Substance Formula (Trivial Name) [CAS Registry Number] InChIKey | $H_s^{cp}$ (at $T^\ominus$) $\left[\dfrac{\mathrm{mol}}{\mathrm{m^3\,Pa}}\right]$ | $\dfrac{\mathrm{d}\ln H_s^{cp}}{\mathrm{d}(1/T)}$ [K] | Reference | Type | Note |
|---|---|---|---|---|---|
| MCM:C23O3MCHO $C_6H_8O_4$ KNKMVXKCHLTCEX-UHFFFAOYSA-N | $1.4\times10^4$ $6.0\times10^3$ $8.1\times10^1$ | | Wang et al. (2017) Wang et al. (2017) Wang et al. (2017) | Q Q Q | 80, 238 80, 239 80, 240 |
| MCM:C5OCO3M $C_6H_8O_4$ MFFVRQLCBYBQSQ-UHFFFAOYSA-N | $1.4\times10^4$ $5.6\times10^3$ $4.8\times10^1$ | | Wang et al. (2017) Wang et al. (2017) Wang et al. (2017) | Q Q Q | 80, 238 80, 239 80, 240 |
| MCM:C23O3ECHO $C_7H_{10}O_4$ RCPFVGJRKWBYKE-UHFFFAOYSA-N | $1.1\times10^4$ $3.2\times10^3$ $4.6\times10^1$ | | Wang et al. (2017) Wang et al. (2017) Wang et al. (2017) | Q Q Q | 80, 238 80, 239 80, 240 |
| MCM:C1014CO $C_{10}H_{16}O_4$ MPJVRMHYAPYGFM-UHFFFAOYSA-N | $4.6\times10^3$ $2.0\times10^4$ $4.8\times10^3$ | | Wang et al. (2017) Wang et al. (2017) Wang et al. (2017) | Q Q Q | 80, 238 80, 239 80, 240 |
| MCM:BCLKBOC $C_{14}H_{22}O_4$ YETGIZICMMLUOK-UHFFFAOYSA-N | $7.8\times10^3$ $3.6\times10^4$ $5.6\times10^4$ | | Wang et al. (2017) Wang et al. (2017) Wang et al. (2017) | Q Q Q | 80, 238 80, 239 80, 240 |
| MCM:C148CO $C_{14}H_{20}O_5$ WKRISEJACIFSSJ-UHFFFAOYSA-N | $5.8\times10^6$ $3.5\times10^5$ $1.5\times10^3$ | | Wang et al. (2017) Wang et al. (2017) Wang et al. (2017) | Q Q Q | 80, 238 80, 239 80, 240 |
| MCM:C148OOH $C_{14}H_{22}O_6$ QNGFZXSUHFZFIE-UHFFFAOYSA-N | $6.8\times10^8$ $3.0\times10^6$ $4.4\times10^4$ | | Wang et al. (2017) Wang et al. (2017) Wang et al. (2017) | Q Q Q | 80, 238 80, 239 80, 240 |
| MCM:C148OH $C_{14}H_{22}O_5$ LXDPLKVHKWYPKP-UHFFFAOYSA-N | $1.3\times10^6$ $7.6\times10^6$ $9.8\times10^4$ | | Wang et al. (2017) Wang et al. (2017) Wang et al. (2017) | Q Q Q | 80, 238 80, 239 80, 240 |
| MCM:ACETC2CO2H $C_5H_8O_4$ RFEXARYJXBYPLD-UHFFFAOYSA-N | $2.9\times10^3$ $4.8\times10^4$ $6.5\times10^4$ | | Wang et al. (2017) Wang et al. (2017) Wang et al. (2017) | Q Q Q | 80, 238 80, 239 80, 240 |
| MCM:C24O3CCO2H $C_5H_6O_5$ HSSKPURIBYAXCO-UHFFFAOYSA-N | $1.6\times10^5$ $3.5\times10^6$ $7.3\times10^4$ | | Wang et al. (2017) Wang et al. (2017) Wang et al. (2017) | Q Q Q | 80, 238 80, 239 80, 240 |
| MCM:IPRACBCO2H $C_5H_8O_4$ WTLNOANVTIKPEE-UHFFFAOYSA-N | $3.5\times10^3$ $6.9\times10^3$ $5.0\times10^3$ | | Wang et al. (2017) Wang et al. (2017) Wang et al. (2017) | Q Q Q | 80, 238 80, 239 80, 240 |
| MCM:MTBEALCO2H $C_5H_8O_4$ BXXOFUQIACXFIW-UHFFFAOYSA-N | $2.8\times10^3$ $2.1\times10^3$ $2.0\times10^3$ | | Wang et al. (2017) Wang et al. (2017) Wang et al. (2017) | Q Q Q | 80, 238 80, 239 80, 240 |
| MCM:C1013CO2H $C_{11}H_{18}O_4$ QWGCLZMSSSTRLX-UHFFFAOYSA-N | $3.0\times10^3$ $3.1\times10^4$ $5.0\times10^4$ | | Wang et al. (2017) Wang et al. (2017) Wang et al. (2017) | Q Q Q | 80, 238 80, 239 80, 240 |





Table A3.8: Esters (RCOOR) (...continued)

| Substance<br>Formula<br>(Trivial Name)<br>[CAS Registry Number]<br>InChIKey | $H_s^{cp}$<br>(at $T^{\ominus}$)<br>$\left[\dfrac{\text{mol}}{\text{m}^3\,\text{Pa}}\right]$ | $\dfrac{\text{d}\ln H_s^{cp}}{\text{d}(1/T)}$<br><br>[K] | Reference | Type | Note |
|---|---|---|---|---|---|
| MCM:CO14O3CO2H | $3.3\times10^6$ | | Wang et al. (2017) | Q | 80, 238 |
| $C_4H_4O_5$ | $5.4\times10^4$ | | Wang et al. (2017) | Q | 80, 239 |
| OPFWUCMVKDVVTF-UHFFFAOYSA-N | $1.7\times10^5$ | | Wang et al. (2017) | Q | 80, 240 |
| MCM:C23O3CCO2H | $2.3\times10^6$ | | Wang et al. (2017) | Q | 80, 238 |
| $C_5H_6O_5$ | $9.1\times10^6$ | | Wang et al. (2017) | Q | 80, 239 |
| WRKCXTWVOWDGBE-UHFFFAOYSA-N | $1.7\times10^5$ | | Wang et al. (2017) | Q | 80, 240 |
| MCM:C23O3MCO2H | $2.0\times10^6$ | | Wang et al. (2017) | Q | 80, 238 |
| $C_6H_8O_5$ | $1.6\times10^6$ | | Wang et al. (2017) | Q | 80, 239 |
| OGSBZWITKXVLIV-UHFFFAOYSA-N | $9.3\times10^4$ | | Wang et al. (2017) | Q | 80, 240 |
| MCM:C23O3ECO2H | $1.7\times10^6$ | | Wang et al. (2017) | Q | 80, 238 |
| $C_7H_{10}O_5$ | $6.0\times10^5$ | | Wang et al. (2017) | Q | 80, 239 |
| OEONINCNUUTEPU-UHFFFAOYSA-N | $2.5\times10^4$ | | Wang et al. (2017) | Q | 80, 240 |



### A3.9   Ethers (ROR)

Table A3.9: Ethers (ROR)

| Substance<br>Formula<br>(Trivial Name)<br>[CAS Registry Number]<br>InChIKey | $H_s^{cp}$<br>(at $T^{\ominus}$)<br>$\left[\dfrac{\mathrm{mol}}{\mathrm{m^3\,Pa}}\right]$ | $\dfrac{\mathrm{d}\ln H_s^{cp}}{\mathrm{d}(1/T)}$<br><br>[K] | Reference | Type | Note |
|---|---|---|---|---|---|
| dimethyl ether | $9.9\times10^{-3}$ | | Duchowicz et al. (2020) | V | 186 |
| $CH_3OCH_3$ | $1.7\times10^{-3}$ | | HSDB (2015) | V | |
| [115-10-6] | $7.6\times10^{-2}$ | | Mackay et al. (2006c) | V | |
| LCGLNKUTAGEVQW-UHFFFAOYSA-N | $1.3\times10^{-1}$ | | Mackay et al. (1993) | V | |
| | $9.9\times10^{-3}$ | | Hine and Mookerjee (1975) | V | |
| | $9.8\times10^{-3}$ | | Hine and Weimar (1965) | R | |
| | $1.0\times10^{-2}$ | 4900 | Bagno et al. (1991) | T | 473 |
| | $1.5\times10^{-2}$ | | Yaws (2003) | X | 237, 28 |
| | $6.5\times10^{-3}$ | | Hayer et al. (2022) | Q | 20 |
| | $5.1\times10^{-2}$ | | Duchowicz et al. (2020) | Q | |
| | $3.0\times10^{-3}$ | | Wang et al. (2017) | Q | 80, 238 |
| | $3.0\times10^{-2}$ | | Wang et al. (2017) | Q | 80, 239 |
| | $1.1\times10^{-2}$ | | Wang et al. (2017) | Q | 80, 240 |
| | $7.1\times10^{-3}$ | | Gharagheizi et al. (2012) | Q | |
| | $1.5\times10^{-2}$ | | Gharagheizi et al. (2010) | Q | 246 |
| | $1.8\times10^{-2}$ | | Hilal et al. (2008) | Q | |
| | $1.4\times10^{-2}$ | | Modarresi et al. (2007) | Q | 67 |
| | $7.5\times10^{-3}$ | | Yaffe et al. (2003) | Q | 248, 249 |
| | $8.4\times10^{-3}$ | | Katritzky et al. (1998) | Q | |
| | $2.2\times10^{-3}$ | | Nirmalakhandan et al. (1997) | Q | |
| | $5.3\times10^{-2}$ | | Russell et al. (1992) | Q | 358 |
| | $1.2\times10^{-2}$ | | Suzuki et al. (1992) | Q | 232 |
| | $1.5\times10^{-2}$ | | Yaws (1999) | ? | 21, 28 |
| | $6.2\times10^{-3}$ | | Abraham and Weathersby (1994) | ? | 21 |
| | $9.9\times10^{-3}$ | | Abraham et al. (1990) | ? | |
| ethyl methyl ether | $8.2\times10^{-3}$ | | Duchowicz et al. (2020) | V | 186 |
| $C_2H_5OCH_3$ | $1.4\times10^{-2}$ | | Bagno et al. (1991) | T | 473 |
| [540-67-0] | $1.9\times10^{-2}$ | | Duchowicz et al. (2020) | Q | |
| XOBKSJJDNFUZPF-UHFFFAOYSA-N | $1.5\times10^{-2}$ | | HSDB (2015) | Q | 99 |
| | $1.5\times10^{-2}$ | | Hilal et al. (2008) | Q | |
| | $3.0\times10^{-2}$ | | English and Carroll (2001) | Q | 230, 231 |
| | $9.7\times10^{-3}$ | | Katritzky et al. (1998) | Q | |
| | $1.9\times10^{-3}$ | | Nirmalakhandan et al. (1997) | Q | |
| | $8.9\times10^{-3}$ | | Saxena and Hildemann (1996) | E | 401 |
| | $1.8\times10^{-2}$ | | Yaws (1999) | ? | 21 |
| diethyl ether | $9.9\times10^{-3}$ | 5800 | Burkholder et al. (2019) | L | 1 |
| $C_2H_5OC_2H_5$ | $2.0\times10^{-2}$ | 5800 | Burkholder et al. (2015) | L | |
| [60-29-7] | $9.9\times10^{-3}$ | 5800 | Brockbank (2013) | L | 1 |
| RTZKZFJDLAIYFH-UHFFFAOYSA-N | $5.0\times10^{-3}$ | | Steward et al. (1973) | L | 14 |
| | $9.6\times10^{-3}$ | 5000 | Allott et al. (1973) | L | |
| | $1.1\times10^{-2}$ | 6600 | Hiatt (2013) | M | |
| | $9.5\times10^{-2}$ | | Helburn et al. (2008) | M | |
| | $1.0\times10^{-2}$ | 5700 | Ondo and Dohnal (2007) | M | 1 |
| | $1.1\times10^{-2}$ | | Nielsen et al. (1994) | M | |



Table A3.9: Ethers (ROR) (...continued)

| Substance<br>Formula<br>(Trivial Name)<br>[CAS Registry Number]<br>InChIKey | $H_s^{cp}$<br>(at $T^{\ominus}$)<br>$\left[\dfrac{\text{mol}}{\text{m}^3\,\text{Pa}}\right]$ | $\dfrac{\text{d}\ln H_s^{cp}}{\text{d}(1/T)}$<br><br>[K] | Reference | Type | Note |
|---|---|---|---|---|---|
| | $3.3\times10^{-3}$ | | Yu (1992) | M | 12 |
| | $7.0\times10^{-3}$ | 3900 | Lamarche and Droste (1989) | M | 345 |
| | $6.3\times10^{-3}$ | | Guitart et al. (1989) | M | 14 |
| | $1.3\times10^{-2}$ | 7400 | Bachofen and Farhi (1971) | M | |
| | $6.3\times10^{-3}$ | | Brody et al. (1971) | M | 14 |
| | $7.8\times10^{-3}$ | | Signer et al. (1969) | M | |
| | $5.1\times10^{-3}$ | | Eger et al. (1963) | M | 14 |
| | $1.1\times10^{-2}$ | | Mackay et al. (2006c) | V | |
| | $1.0\times10^{-2}$ | 5800 | Fukuchi et al. (2002) | V | |
| | $1.1\times10^{-2}$ | | Mackay et al. (1993) | V | |
| | $8.7\times10^{-3}$ | | Hwang et al. (1992) | V | |
| | $1.1\times10^{-2}$ | | Hine and Weimar (1965) | V | |
| | $1.1\times10^{-2}$ | | Butler and Ramchandani (1935) | V | |
| | $6.0\times10^{-3}$ | 5700 | Bagno et al. (1991) | T | 473 |
| | $1.2\times10^{-2}$ | | Yaws (2003) | X | 237 |
| | $4.3\times10^{-3}$ | | Keshavarz et al. (2022) | Q | |
| | $6.8\times10^{-3}$ | | Duchowicz et al. (2020) | Q | 184 |
| | $2.1\times10^{-3}$ | | Wang et al. (2017) | Q | 80, 238 |
| | $1.6\times10^{-2}$ | | Wang et al. (2017) | Q | 80, 239 |
| | $2.2\times10^{-3}$ | | Wang et al. (2017) | Q | 80, 240 |
| | $7.7\times10^{-3}$ | | Li et al. (2014) | Q | 241 |
| | $1.3\times10^{-2}$ | | Gharagheizi et al. (2012) | Q | |
| | $6.2\times10^{-3}$ | | Raventos-Duran et al. (2010) | Q | 242, 243 |
| | $1.2\times10^{-2}$ | | Raventos-Duran et al. (2010) | Q | 244 |
| | $6.2\times10^{-3}$ | | Raventos-Duran et al. (2010) | Q | 245 |
| | $1.1\times10^{-2}$ | | Gharagheizi et al. (2010) | Q | 246 |
| | $7.0\times10^{-3}$ | | Hilal et al. (2008) | Q | |
| | $1.3\times10^{-2}$ | | Modarresi et al. (2007) | Q | 67 |
| | | 5300 | Kühne et al. (2005) | Q | |
| | $8.6\times10^{-3}$ | | Yaffe et al. (2003) | Q | 248, 249 |
| | $2.4\times10^{-2}$ | | English and Carroll (2001) | Q | 230, 231 |
| | $1.2\times10^{-2}$ | | Katritzky et al. (1998) | Q | |
| | $1.7\times10^{-3}$ | | Nirmalakhandan et al. (1997) | Q | |
| | $4.6\times10^{-2}$ | | Russell et al. (1992) | Q | 279 |
| | $7.3\times10^{-3}$ | | Suzuki et al. (1992) | Q | 232 |
| | $8.0\times10^{-3}$ | | Duchowicz et al. (2020) | ? | 185, 21 |
| | | 5700 | Kühne et al. (2005) | ? | |
| | $1.2\times10^{-2}$ | | Yaws (1999) | ? | 21 |
| | $5.8\times10^{-3}$ | | Abraham and Weathersby (1994) | ? | 21 |
| | $7.7\times10^{-3}$ | | Hoff et al. (1993) | ? | 21 |
| | $6.0\times10^{-3}$ | | Abraham et al. (1990) | ? | |
| diethyl ether-d10<br>$C_2D_5OC_2D_5$<br>[2679-89-2]<br>RTZKZFJDLAIYFH-MWUKXHIBSA-N | $1.3\times10^{-2}$ | 6500 | Hiatt (2013) | M | |



Table A3.9: Ethers (ROR) (...continued)

| Substance Formula (Trivial Name) [CAS Registry Number] InChIKey | $H_s^{cp}$ (at $T^\ominus$) $\left[\dfrac{\mathrm{mol}}{\mathrm{m^3\,Pa}}\right]$ | $\dfrac{\mathrm{d}\ln H_s^{cp}}{\mathrm{d}(1/T)}$ [K] | Reference | Type | Note |
|---|---|---|---|---|---|
| methyl propyl ether | $6.6\times10^{-3}$ | | Duchowicz et al. (2020) | V | 186 |
| $CH_3OC_3H_7$ | $6.7\times10^{-3}$ | | Meylan and Howard (1991) | V | |
| [557-17-5] | $6.7\times10^{-3}$ | | Hine and Mookerjee (1975) | V | |
| VNKYTQGIUYNRMY-UHFFFAOYSA-N | $6.9\times10^{-3}$ | | Yaws (2003) | X | 237 |
| | $2.0\times10^{-2}$ | | Duchowicz et al. (2020) | Q | |
| | $5.2\times10^{-3}$ | | Gharagheizi et al. (2012) | Q | |
| | $6.2\times10^{-3}$ | | Raventos-Duran et al. (2010) | Q | 271, 243 |
| | $1.6\times10^{-2}$ | | Raventos-Duran et al. (2010) | Q | 244 |
| | $6.2\times10^{-3}$ | | Raventos-Duran et al. (2010) | Q | 245 |
| | $8.9\times10^{-3}$ | | Gharagheizi et al. (2010) | Q | 246 |
| | $1.1\times10^{-2}$ | | Hilal et al. (2008) | Q | |
| | $6.9\times10^{-3}$ | | Yaffe et al. (2003) | Q | 248, 249 |
| | $1.1\times10^{-2}$ | | Katritzky et al. (1998) | Q | |
| | $1.5\times10^{-3}$ | | Nirmalakhandan et al. (1997) | Q | |
| | $7.3\times10^{-3}$ | | Suzuki et al. (1992) | Q | 232 |
| | $6.5\times10^{-3}$ | | Meylan and Howard (1991) | Q | |
| | $6.9\times10^{-3}$ | | Yaws (1999) | ? | 21 |
| methyl 2-propyl ether | $1.1\times10^{-2}$ | | Duchowicz et al. (2020) | V | 186 |
| $CH_3OC_3H_7$ | $1.2\times10^{-2}$ | | Hine and Mookerjee (1975) | V | |
| (methyl isopropyl ether) | $1.2\times10^{-2}$ | | Yaws (2003) | X | 258 |
| [598-53-8] | $1.1\times10^{-2}$ | | Yaws (2003) | X | 237 |
| RMGHERXMTMUMMV-UHFFFAOYSA-N | $7.1\times10^{-3}$ | | Dupeux et al. (2022) | Q | 259 |
| | $7.8\times10^{-3}$ | | Duchowicz et al. (2020) | Q | |
| | $1.4\times10^{-2}$ | | Gharagheizi et al. (2012) | Q | |
| | $6.2\times10^{-3}$ | | Raventos-Duran et al. (2010) | Q | 242, 243 |
| | $1.2\times10^{-2}$ | | Raventos-Duran et al. (2010) | Q | 244 |
| | $6.2\times10^{-3}$ | | Raventos-Duran et al. (2010) | Q | 245 |
| | $1.2\times10^{-2}$ | | Gharagheizi et al. (2010) | Q | 246 |
| | $8.2\times10^{-3}$ | | Hilal et al. (2008) | Q | |
| | $1.2\times10^{-2}$ | | Yaffe et al. (2003) | Q | 248, 249 |
| | $1.1\times10^{-2}$ | | Katritzky et al. (1998) | Q | |
| | $6.5\times10^{-3}$ | | Suzuki et al. (1992) | Q | 232 |
| | $1.1\times10^{-2}$ | | Yaws (1999) | ? | 21 |
| divinyl ether | $5.4\times10^{-4}$ | | Steward et al. (1973) | L | 14 |
| $C_4H_6O$ | $5.4\times10^{-4}$ | | Allott et al. (1973) | L | 14 |
| [109-93-3] | $1.2\times10^{-3}$ | | Duchowicz et al. (2020) | V | 186 |
| QYKIQEUNHZKYBP-UHFFFAOYSA-N | $5.6\times10^{-4}$ | | Yaws (2003) | X | 237, 14 |
| | $7.3\times10^{-2}$ | | Duchowicz et al. (2020) | Q | |
| | $7.8\times10^{-3}$ | | Raventos-Duran et al. (2010) | Q | 242, 243 |
| | $3.9\times10^{-4}$ | | Raventos-Duran et al. (2010) | Q | 244 |
| | $2.5\times10^{-4}$ | | Raventos-Duran et al. (2010) | Q | 245 |
| | $6.0\times10^{-4}$ | | Gharagheizi et al. (2010) | Q | 246 |
| | $3.8\times10^{-4}$ | | Hilal et al. (2008) | Q | |
| | $2.0\times10^{-3}$ | | Nirmalakhandan et al. (1997) | Q | |
| | $5.8\times10^{-4}$ | | Abraham and Weathersby (1994) | ? | 21 |



Table A3.9: Ethers (ROR) (...continued)

| Substance Formula (Trivial Name) [CAS Registry Number] InChIKey | $H_s^{cp}$ (at $T^{\ominus}$) $\left[\dfrac{\text{mol}}{\text{m}^3\,\text{Pa}}\right]$ | $\dfrac{\mathrm{d}\ln H_s^{cp}}{\mathrm{d}(1/T)}$ [K] | Reference | Type | Note |
|---|---|---|---|---|---|
| methyl butyl ether | $6.6\times10^{-3}$ | 6100 | Brockbank (2013) | L | 1, 518 |
| $C_5H_{12}O$ | $5.5\times10^{-3}$ | | Duchowicz et al. (2020) | V | 186 |
| [628-28-4] | $4.4\times10^{-3}$ | | Amoore and Buttery (1978) | V | |
| CXBDYQVECUFKRK-UHFFFAOYSA-N | $7.1\times10^{-3}$ | | Yaws (2003) | X | 237 |
| | $2.1\times10^{-2}$ | | Duchowicz et al. (2020) | Q | |
| | $4.6\times10^{-3}$ | | Gharagheizi et al. (2012) | Q | |
| | $4.9\times10^{-3}$ | | Raventos-Duran et al. (2010) | Q | 242, 243 |
| | $1.2\times10^{-2}$ | | Raventos-Duran et al. (2010) | Q | 244 |
| | $4.9\times10^{-3}$ | | Raventos-Duran et al. (2010) | Q | 245 |
| | $5.3\times10^{-3}$ | | Gharagheizi et al. (2010) | Q | 246 |
| | $9.6\times10^{-3}$ | | Modarresi et al. (2007) | Q | 67 |
| | $5.6\times10^{-3}$ | | Yaffe et al. (2003) | Q | 248, 249 |
| | $1.1\times10^{-2}$ | | Katritzky et al. (1998) | Q | |
| 2-methoxybutane | $6.7\times10^{-3}$ | | Duchowicz et al. (2020) | V | 186 |
| $C_5H_{12}O$ | $7.1\times10^{-3}$ | | Yaws (2003) | X | 237 |
| (methyl *sec*-butyl ether) | $8.1\times10^{-3}$ | | Duchowicz et al. (2020) | Q | |
| [6795-87-5] | $7.3\times10^{-3}$ | | Gharagheizi et al. (2012) | Q | |
| FVNIMHIOIXPIQT-UHFFFAOYSA-N | $4.9\times10^{-3}$ | | Raventos-Duran et al. (2010) | Q | 242, 243 |
| | $9.9\times10^{-3}$ | | Raventos-Duran et al. (2010) | Q | 244 |
| | $4.9\times10^{-3}$ | | Raventos-Duran et al. (2010) | Q | 245 |
| | $7.2\times10^{-3}$ | | Gharagheizi et al. (2010) | Q | 246 |
| | $6.2\times10^{-3}$ | | Hilal et al. (2008) | Q | |
| | $1.6\times10^{-2}$ | | Modarresi et al. (2007) | Q | 67 |
| | $8.6\times10^{-3}$ | | Yaws (1999) | ? | 21 |
| methyl isobutyl ether | $4.3\times10^{-3}$ | 6700 | Brockbank (2013) | L | 1 |
| $C_5H_{12}O$ | $4.5\times10^{-3}$ | | Duchowicz et al. (2020) | V | 186 |
| [625-44-5] | $7.2\times10^{-3}$ | | Yaws (2003) | X | 237 |
| ZYVYEJXMYBUCMN-UHFFFAOYSA-N | $8.1\times10^{-3}$ | | Duchowicz et al. (2020) | Q | |
| | $2.3\times10^{-3}$ | | Gharagheizi et al. (2012) | Q | |
| | $6.9\times10^{-3}$ | | Gharagheizi et al. (2010) | Q | 246 |
| | $7.1\times10^{-3}$ | | Modarresi et al. (2007) | Q | 67 |
| | $5.6\times10^{-3}$ | | Yaffe et al. (2003) | Q | 248, 272 |
| | $1.2\times10^{-2}$ | | Katritzky et al. (1998) | Q | |
| | $8.6\times10^{-3}$ | | Yaws (1999) | ? | 21 |
| methyl *tert*-butyl ether | $1.2\times10^{-2}$ | 5100 | Schwardt et al. (2021) | L | 1 |
| $CH_3OC(CH_3)_3$ | $1.3\times10^{-2}$ | 5900 | Brockbank (2013) | L | 1 |
| (MTBE) | $1.7\times10^{-2}$ | 9100 | Hiatt (2013) | M | |
| [1634-04-4] | $2.4\times10^{-2}$ | 18000 | Zhang et al. (2013) | M | 324 |
| BZLVMXJERCGZMT-UHFFFAOYSA-N | $3.2\times10^{-2}$ | | Zhang et al. (2013) | M | 325 |
| | $1.9\times10^{-2}$ | 5300 | Hwang et al. (2010) | M | 33, 519, 11 |
| | $1.1\times10^{-2}$ | 4800 | Sieg et al. (2009) | M | 326 |
| | $1.1\times10^{-2}$ | 4400 | Falabella and Teja (2008) | M | 11, 338 |
| | $1.5\times10^{-2}$ | 5900 | Böhme et al. (2008) | M | |
| | $1.2\times10^{-2}$ | 5100 | Haimi et al. (2006) | M | 520 |
| | $1.2\times10^{-2}$ | 5000 | Arp and Schmidt (2004) | M | 521 |



Table A3.9: Ethers (ROR) (...continued)

| Substance Formula (Trivial Name) [CAS Registry Number] InChIKey | $H_s^{cp}$ (at $T^{\ominus}$) $\left[\dfrac{\text{mol}}{\text{m}^3\,\text{Pa}}\right]$ | $\dfrac{\text{d}\ln H_s^{cp}}{\text{d}(1/T)}$ [K] | Reference | Type | Note |
|---|---|---|---|---|---|
| | $1.4\times10^{-2}$ | 4500 | Fischer et al. (2004) | M | |
| | $7.2\times10^{-3}$ | 3200 | Bierwagen and Keller (2001) | M | |
| | $1.7\times10^{-2}$ | | Miller and Stuart (2000) | M | 72 |
| | $2.3\times10^{-2}$ | | Park et al. (1997) | M | |
| | $1.9\times10^{-2}$ | 15000 | Robbins et al. (1993) | M | 522 |
| | $1.4\times10^{-2}$ | | Mackay et al. (2006c) | V | |
| | $1.0\times10^{-1}$ | 3700 | Fukuchi et al. (2002) | V | |
| | $1.6\times10^{-2}$ | | Park et al. (1997) | V | |
| | $1.4\times10^{-2}$ | | Mackay et al. (1993) | V | |
| | $2.0\times10^{-2}$ | | Hwang et al. (1992) | V | |
| | $1.7\times10^{-2}$ | | Guthrie (1973) | V | |
| | $1.7\times10^{-2}$ | | Bagno et al. (1991) | T | 473 |
| | $1.8\times10^{-2}$ | | Yaws (2003) | X | 258 |
| | $1.0\times10^{-2}$ | | Dupeux et al. (2022) | Q | 259 |
| | $5.7\times10^{-3}$ | | Keshavarz et al. (2022) | Q | |
| | $3.5\times10^{-3}$ | | Duchowicz et al. (2020) | Q | 184 |
| | $1.3\times10^{-3}$ | | Wang et al. (2017) | Q | 80, 238 |
| | $6.0\times10^{-3}$ | | Wang et al. (2017) | Q | 80, 239 |
| | $1.5\times10^{-2}$ | | Wang et al. (2017) | Q | 80, 240 |
| | $2.1\times10^{-2}$ | | Gharagheizi et al. (2012) | Q | |
| | $4.9\times10^{-3}$ | | Raventos-Duran et al. (2010) | Q | 242, 243 |
| | $6.2\times10^{-3}$ | | Raventos-Duran et al. (2010) | Q | 244 |
| | $4.9\times10^{-3}$ | | Raventos-Duran et al. (2010) | Q | 245 |
| | $3.9\times10^{-3}$ | | Hilal et al. (2008) | Q | |
| | $1.3\times10^{-2}$ | | Modarresi et al. (2007) | Q | 67 |
| | | 6300 | Kühne et al. (2005) | Q | |
| | $1.8\times10^{-2}$ | | Yaffe et al. (2003) | Q | 248, 249 |
| | $1.4\times10^{-2}$ | | English and Carroll (2001) | Q | 230, 231 |
| | $1.1\times10^{-2}$ | | Katritzky et al. (1998) | Q | |
| | $8.6\times10^{-4}$ | | Nirmalakhandan et al. (1997) | Q | |
| | $4.3\times10^{-3}$ | | Suzuki et al. (1992) | Q | 232 |
| | $1.7\times10^{-2}$ | | Duchowicz et al. (2020) | ? | 185, 21 |
| | | 6000 | Kühne et al. (2005) | ? | |
| | $1.8\times10^{-2}$ | | Yaws (1999) | ? | 21 |
| ethyl propyl ether $C_2H_5OC_3H_7$ [628-32-0] NVJUHMXYKCUMQA-UHFFFAOYSA-N | $8.7\times10^{-3}$ | | Duchowicz et al. (2020) | V | 186 |
| | $8.6\times10^{-3}$ | | Hine and Mookerjee (1975) | V | |
| | $8.6\times10^{-3}$ | | Butler and Ramchandani (1935) | V | |
| | $6.4\times10^{-3}$ | | Yaws (2003) | X | 237 |
| | $7.7\times10^{-3}$ | | Howard and Meylan (1997) | X | 446 |
| | $7.0\times10^{-3}$ | | Duchowicz et al. (2020) | Q | |
| | $8.2\times10^{-3}$ | | Gharagheizi et al. (2012) | Q | |
| | $4.9\times10^{-3}$ | | Raventos-Duran et al. (2010) | Q | 242, 243 |
| | $9.9\times10^{-3}$ | | Raventos-Duran et al. (2010) | Q | 244 |
| | $4.9\times10^{-3}$ | | Raventos-Duran et al. (2010) | Q | 245 |
| | $7.0\times10^{-3}$ | | Gharagheizi et al. (2010) | Q | 246 |
| | $7.9\times10^{-3}$ | | Hilal et al. (2008) | Q | |
| | $1.4\times10^{-2}$ | | Modarresi et al. (2007) | Q | 67 |



Table A3.9: Ethers (ROR) (...continued)

| Substance Formula (Trivial Name) [CAS Registry Number] InChIKey | $H_s^{cp}$ (at $T^\ominus$) $\left[\dfrac{\mathrm{mol}}{\mathrm{m}^3\,\mathrm{Pa}}\right]$ | $\dfrac{\mathrm{d}\ln H_s^{cp}}{\mathrm{d}(1/T)}$ [K] | Reference | Type | Note |
|---|---|---|---|---|---|
| | $8.6\times10^{-3}$ | | Yaffe et al. (2003) | Q | 248, 249 |
| | $1.2\times10^{-2}$ | | Katritzky et al. (1998) | Q | |
| | $5.7\times10^{-3}$ | | Suzuki et al. (1992) | Q | 232 |
| | $7.9\times10^{-3}$ | | Yaws (1999) | ? | 21 |
| ethyl isopropyl ether $C_5H_{12}O$ [625-54-7] XSJVWZAETSBXKU-UHFFFAOYSA-N | $8.1\times10^{-3}$ $2.0\times10^{-2}$ $9.1\times10^{-3}$ | | Yaws (2003) Gharagheizi et al. (2012) Gharagheizi et al. (2010) | X Q Q | 237 246 |
| 2,2-dimethoxypropane $C_5H_{12}O_2$ [77-76-9] HEWZVZIVELJPQZ-UHFFFAOYSA-N | $1.4\times10^{-1}$ | | Ebert et al. (2023) | ? | 316 |
| methyl 1,2-dimethylpropyl ether $C_6H_{14}O$ [62016-49-3] JPUDLQKLSRSRGN-UHFFFAOYSA-N | $4.6\times10^{-3}$ $3.8\times10^{-3}$ $5.7\times10^{-3}$ | | Yaws (2003) Gharagheizi et al. (2012) Gharagheizi et al. (2010) | X Q Q | 237 246 |
| methyl 1-ethylpropyl ether $C_6H_{14}O$ [36839-67-5] CQRFEDVNTJTKFU-UHFFFAOYSA-N | $4.4\times10^{-3}$ $4.3\times10^{-3}$ $4.2\times10^{-3}$ | | Yaws (2003) Gharagheizi et al. (2012) Gharagheizi et al. (2010) | X Q Q | 237 246 |
| methyl 1-methylbutyl ether $C_6H_{14}O$ [6795-88-6] XSAJCGUYMQTAHL-UHFFFAOYSA-N | $4.2\times10^{-3}$ $6.9\times10^{-3}$ $4.0\times10^{-3}$ | | Yaws (2003) Gharagheizi et al. (2012) Gharagheizi et al. (2010) | X Q Q | 237 246 |
| methyl 2,2-dimethylpropyl ether $C_6H_{14}O$ [1118-00-9] JILHZKWLEAKYRC-UHFFFAOYSA-N | $4.5\times10^{-3}$ $1.2\times10^{-3}$ $4.1\times10^{-3}$ | | Yaws (2003) Gharagheizi et al. (2012) Gharagheizi et al. (2010) | X Q Q | 237 246 |
| methyl 2-methylbutyl ether $C_6H_{14}O$ [62016-48-2] XGLHRCWEOMNVKS-UHFFFAOYSA-N | $4.3\times10^{-3}$ $2.1\times10^{-3}$ $3.9\times10^{-3}$ | | Yaws (2003) Gharagheizi et al. (2012) Gharagheizi et al. (2010) | X Q Q | 237 246 |
| methyl 3-methylbutyl ether $C_6H_{14}O$ [626-91-5] ZQAYBCWERYRAMF-UHFFFAOYSA-N | $4.3\times10^{-3}$ $3.1\times10^{-3}$ $3.9\times10^{-3}$ | | Yaws (2003) Gharagheizi et al. (2012) Gharagheizi et al. (2010) | X Q Q | 237 246 |
| methyl pentyl ether $C_6H_{14}O$ [628-80-8] DBUJFULDVAZULB-UHFFFAOYSA-N | $3.8\times10^{-3}$ $3.9\times10^{-3}$ $3.0\times10^{-3}$ | | Yaws (2003) Gharagheizi et al. (2012) Gharagheizi et al. (2010) | X Q Q | 237 246 |





658                                    Rolf Sander: Compilation of Henry's law constants

Table A3.9: Ethers (ROR) (... continued)

| Substance Formula (Trivial Name) [CAS Registry Number] InChIKey | $H_s^{cp}$ (at $T^\ominus$) $\left[\dfrac{\text{mol}}{\text{m}^3\,\text{Pa}}\right]$ | $\dfrac{\text{d}\ln H_s^{cp}}{\text{d}(1/T)}$ [K] | Reference | Type | Note |
|---|---|---|---|---|---|
| 2-methoxy-2-methylbutane | $9.7\times10^{-3}$ | 6600 | Brockbank (2013) | L | 1 |
| $C_6H_{14}O$ | $5.3\times10^{-2}$ | 9400 | Hwang et al. (2010) | M | 519, 11 |
| (*tert*-amyl methyl ether) | $1.0\times10^{-2}$ | 7000 | Haimi et al. (2006) | M | 523 |
| [994-05-8] | $8.6\times10^{-3}$ | 6500 | Arp and Schmidt (2004) | M | |
| HVZJRWJGKQPSFL-UHFFFAOYSA-N | $5.2\times10^{-3}$ | | Miller and Stuart (2000) | M | 72 |
| | $1.0\times10^{-2}$ | | Dohnal and Hovorka (1999) | M | |
| | $7.0\times10^{-3}$ | | Park et al. (1997) | M | |
| | $8.1\times10^{-3}$ | | Park et al. (1997) | V | |
| | $4.2\times10^{-3}$ | | Yaws (2003) | X | 237 |
| | $3.9\times10^{-3}$ | | Yaws (2003) | X | 237 |
| | $4.7\times10^{-2}$ | | Keshavarz et al. (2022) | Q | |
| | $3.6\times10^{-3}$ | | Duchowicz et al. (2020) | Q | 299 |
| | $1.3\times10^{-2}$ | | Gharagheizi et al. (2012) | Q | |
| | $3.8\times10^{-3}$ | | Gharagheizi et al. (2010) | Q | 246 |
| | $3.8\times10^{-3}$ | | Gharagheizi et al. (2010) | Q | 246 |
| | | 6600 | Kühne et al. (2005) | Q | |
| | $7.5\times10^{-3}$ | | Duchowicz et al. (2020) | ? | 185, 21 |
| | | 6900 | Kühne et al. (2005) | ? | |
| | $5.4\times10^{-3}$ | | Yaws (1999) | ? | 21 |
| | $5.0\times10^{-3}$ | 7600 | Pankow et al. (1996) | ? | |
| 1-ethoxy-butane | $7.6\times10^{-3}$ | 5700 | Brockbank (2013) | L | 1 |
| $C_6H_{14}O$ | $6.4\times10^{-3}$ | | Miller and Stuart (2000) | M | 72 |
| (ethyl butyl ether) | $7.8\times10^{-3}$ | | Mackay et al. (2006c) | V | |
| [628-81-9] | $7.8\times10^{-3}$ | | Mackay et al. (1993) | V | |
| PZHIWRCQKBBTOW-UHFFFAOYSA-N | $4.0\times10^{-3}$ | | Yaws (2003) | X | 237 |
| | $6.7\times10^{-3}$ | | Gharagheizi et al. (2012) | Q | |
| | $4.0\times10^{-3}$ | | Gharagheizi et al. (2010) | Q | 246 |
| | | 5900 | Kühne et al. (2005) | Q | |
| | | 5000 | Kühne et al. (2005) | ? | |
| ethyl isobutyl ether | $4.9\times10^{-3}$ | | Yaws (2003) | X | 237 |
| $C_6H_{14}O$ | $3.5\times10^{-3}$ | | Gharagheizi et al. (2012) | Q | |
| [627-02-1] | $4.9\times10^{-3}$ | | Gharagheizi et al. (2010) | Q | 246 |
| RQUBQBFVDOLUKC-UHFFFAOYSA-N | | | | | |
| ethyl *sec*-butyl ether | $4.9\times10^{-3}$ | | Yaws (2003) | X | 237 |
| $C_6H_{14}O$ | $1.1\times10^{-2}$ | | Gharagheizi et al. (2012) | Q | |
| [2679-87-0] | $5.3\times10^{-3}$ | | Gharagheizi et al. (2010) | Q | 246 |
| VSCUCHUDCLERMY-UHFFFAOYSA-N | | | | | |
| ethyl *tert*-butyl ether | $7.2\times10^{-3}$ | 6900 | Brockbank (2013) | L | 1 |
| $C_2H_5OC(CH_3)_3$ | $1.2\times10^{-1}$ | 13000 | Hwang et al. (2010) | M | 519, 11 |
| (ETBE) | $6.3\times10^{-3}$ | 6600 | Sieg et al. (2009) | M | 326 |
| [637-92-3] | $4.4\times10^{-3}$ | 4300 | Falabella and Teja (2008) | M | 11, 338 |
| NUMQCACRALPSHD-UHFFFAOYSA-N | $6.4\times10^{-3}$ | 7300 | Haimi et al. (2006) | M | 524 |
| | $6.1\times10^{-3}$ | 6500 | Arp and Schmidt (2004) | M | |
| | $4.2\times10^{-3}$ | | Miller and Stuart (2000) | M | 72 |
| | $5.2\times10^{-3}$ | | Yaws (2003) | X | 237 |



Table A3.9: Ethers (ROR) (...continued)

| Substance Formula (Trivial Name) [CAS Registry Number] InChIKey | $H_s^{cp}$ (at $T^\ominus$) $\left[\dfrac{\text{mol}}{\text{m}^3\,\text{Pa}}\right]$ | $\dfrac{\text{d}\ln H_s^{cp}}{\text{d}(1/T)}$ [K] | Reference | Type | Note |
|---|---|---|---|---|---|
| | $7.8\times10^{-3}$ | | Keshavarz et al. (2022) | Q | |
| | $1.2\times10^{-3}$ | | Duchowicz et al. (2020) | Q | 299 |
| | $1.1\times10^{-3}$ | | Wang et al. (2017) | Q | 80, 238 |
| | $4.1\times10^{-3}$ | | Wang et al. (2017) | Q | 80, 239 |
| | $1.5\times10^{-2}$ | | Wang et al. (2017) | Q | 80, 240 |
| | $2.9\times10^{-2}$ | | Gharagheizi et al. (2012) | Q | |
| | $5.2\times10^{-3}$ | | Gharagheizi et al. (2010) | Q | 246 |
| | $1.2\times10^{-2}$ | | Katritzky et al. (1998) | Q | |
| | $6.0\times10^{-3}$ | | Duchowicz et al. (2020) | ? | 185, 21 |
| | $3.7\times10^{-3}$ | 7600 | Pankow et al. (1996) | ? | |
| dipropyl ether $C_3H_7OC_3H_7$ [111-43-3] POLCUAVZOMRGSN-UHFFFAOYSA-N | $2.9\times10^{-3}$ | 6400 | Brockbank (2013) | L | 1 |
| | $3.0\times10^{-3}$ | | Li and Carr (1993) | M | |
| | $2.9\times10^{-3}$ | | Li et al. (1993) | M | |
| | $2.2\times10^{-3}$ | 9100 | Hartkopf and Karger (1973) | M | |
| | $3.9\times10^{-3}$ | | Mackay et al. (2006c) | V | |
| | $3.9\times10^{-3}$ | | Mackay et al. (1993) | V | |
| | $5.7\times10^{-3}$ | | Hwang et al. (1992) | V | |
| | $2.9\times10^{-3}$ | | Hine and Mookerjee (1975) | V | |
| | $2.8\times10^{-3}$ | | Butler and Ramchandani (1935) | V | |
| | $4.5\times10^{-3}$ | | Yaws (2003) | X | 258 |
| | $4.5\times10^{-3}$ | | Yaws (2003) | X | 237 |
| | $1.2\times10^{-3}$ | | Dupeux et al. (2022) | Q | 259 |
| | $7.8\times10^{-3}$ | | Keshavarz et al. (2022) | Q | |
| | $7.2\times10^{-3}$ | | Duchowicz et al. (2020) | Q | 184 |
| | $4.4\times10^{-3}$ | | Gharagheizi et al. (2012) | Q | |
| | $3.9\times10^{-3}$ | | Raventos-Duran et al. (2010) | Q | 242, 243 |
| | $7.8\times10^{-3}$ | | Raventos-Duran et al. (2010) | Q | 244 |
| | $3.9\times10^{-3}$ | | Raventos-Duran et al. (2010) | Q | 245 |
| | $4.1\times10^{-3}$ | | Gharagheizi et al. (2010) | Q | 246 |
| | $6.0\times10^{-3}$ | | Hilal et al. (2008) | Q | |
| | $7.5\times10^{-3}$ | | Modarresi et al. (2007) | Q | 67 |
| | | 5900 | Kühne et al. (2005) | Q | |
| | $4.5\times10^{-3}$ | | Yaffe et al. (2003) | Q | 248, 249 |
| | $1.1\times10^{-2}$ | | Katritzky et al. (1998) | Q | |
| | $1.0\times10^{-3}$ | | Nirmalakhandan et al. (1997) | Q | |
| | $4.3\times10^{-3}$ | | Suzuki et al. (1992) | Q | 232 |
| | $4.5\times10^{-3}$ | | Duchowicz et al. (2020) | ? | 185, 21 |
| | | 7300 | Kühne et al. (2005) | ? | |
| | $4.5\times10^{-3}$ | | Yaws (1999) | ? | 21 |
| | $1.9\times10^{-3}$ | | Hoff et al. (1993) | ? | 21 |
| | $4.5\times10^{-3}$ | | Yaws and Yang (1992) | ? | 21 |
| | $2.9\times10^{-3}$ | | Abraham et al. (1990) | ? | |



Table A3.9: Ethers (ROR) (. . . continued)

| Substance Formula (Trivial Name) [CAS Registry Number] InChIKey | $H_s^{cp}$ (at $T^\ominus$) $\left[\dfrac{\mathrm{mol}}{\mathrm{m^3\,Pa}}\right]$ | $\dfrac{\mathrm{d}\ln H_s^{cp}}{\mathrm{d}(1/T)}$ [K] | Reference | Type | Note |
|---|---|---|---|---|---|
| diisopropyl ether | $4.5\times10^{-3}$ | 6600 | Brockbank (2013) | L | 1 |
| $C_3H_7OC_3H_7$ | $1.5\times10^{-2}$ | 8400 | Hwang et al. (2010) | M | 519, 11 |
| [108-20-3] | $3.9\times10^{-3}$ | 6400 | Arp and Schmidt (2004) | M | |
| ZAFNJMIOTHYJRJ-UHFFFAOYSA-N | $4.3\times10^{-3}$ | | Miller and Stuart (2000) | M | 72 |
| | $4.7\times10^{-3}$ | | Dohnal and Hovorka (1999) | M | |
| | $4.8\times10^{-3}$ | | Nielsen et al. (1994) | M | |
| | $4.2\times10^{-3}$ | | Li and Carr (1993) | M | |
| | $4.4\times10^{-3}$ | | Li et al. (1993) | M | |
| | $2.8\times10^{-3}$ | | Guitart et al. (1989) | M | 14 |
| | $4.3\times10^{-3}$ | | HSDB (2015) | V | |
| | $3.9\times10^{-3}$ | | Mackay et al. (2006c) | V | |
| | $4.8\times10^{-3}$ | 6200 | Fukuchi et al. (2002) | V | |
| | $3.1\times10^{-3}$ | 6400 | Pankow et al. (1996) | V | |
| | $3.9\times10^{-3}$ | | Mackay et al. (1993) | V | |
| | $9.9\times10^{-4}$ | | Hine and Mookerjee (1975) | V | |
| | $9.8\times10^{-4}$ | | Hine and Weimar (1965) | V | |
| | $5.7\times10^{-3}$ | | Yaws (2003) | X | 258 |
| | $5.6\times10^{-3}$ | | Yaws (2003) | X | 237 |
| | $5.7\times10^{-3}$ | | Dupeux et al. (2022) | Q | 259 |
| | $7.8\times10^{-3}$ | | Keshavarz et al. (2022) | Q | |
| | $1.1\times10^{-3}$ | | Duchowicz et al. (2020) | Q | |
| | $1.8\times10^{-3}$ | | Wang et al. (2017) | Q | 80, 238 |
| | $6.2\times10^{-3}$ | | Wang et al. (2017) | Q | 80, 239 |
| | $1.2\times10^{-2}$ | | Wang et al. (2017) | Q | 80, 240 |
| | $2.2\times10^{-2}$ | | Gharagheizi et al. (2012) | Q | |
| | $3.9\times10^{-3}$ | | Raventos-Duran et al. (2010) | Q | 242, 243 |
| | $6.2\times10^{-3}$ | | Raventos-Duran et al. (2010) | Q | 244 |
| | $3.9\times10^{-3}$ | | Raventos-Duran et al. (2010) | Q | 245 |
| | $5.5\times10^{-3}$ | | Gharagheizi et al. (2010) | Q | 246 |
| | $3.7\times10^{-3}$ | | Hilal et al. (2008) | Q | |
| | $9.6\times10^{-3}$ | | Modarresi et al. (2007) | Q | 67 |
| | | 6600 | Kühne et al. (2005) | Q | |
| | $4.5\times10^{-3}$ | | Yaffe et al. (2003) | Q | 248, 249 |
| | $1.2\times10^{-2}$ | | Katritzky et al. (1998) | Q | |
| | $8.0\times10^{-4}$ | | Nirmalakhandan et al. (1997) | Q | |
| | $3.5\times10^{-3}$ | | Suzuki et al. (1992) | Q | 232 |
| | $3.9\times10^{-3}$ | | Duchowicz et al. (2020) | ? | 185, 21 |
| | | 7200 | Kühne et al. (2005) | ? | |
| | $5.6\times10^{-3}$ | | Yaws (1999) | ? | 21 |
| | $5.7\times10^{-3}$ | | Yaws and Yang (1992) | ? | 21 |
| | $9.9\times10^{-4}$ | | Abraham et al. (1990) | ? | |



Table A3.9: Ethers (ROR) (...continued)

| Substance Formula (Trivial Name) [CAS Registry Number] InChIKey | $H_s^{cp}$ (at $T^{\ominus}$) $\left[\dfrac{\mathrm{mol}}{\mathrm{m}^3\,\mathrm{Pa}}\right]$ | $\dfrac{\mathrm{d}\ln H_s^{cp}}{\mathrm{d}(1/T)}$ [K] | Reference | Type | Note |
|---|---|---|---|---|---|
| propyl isopropyl ether | $5.0\times10^{-3}$ | | Yaws (2003) | X | 237 |
| $C_6H_{14}O$ | $1.1\times10^{-2}$ | | Gharagheizi et al. (2012) | Q | |
| [627-08-7] | $3.9\times10^{-3}$ | | Raventos-Duran et al. (2010) | Q | 242, 243 |
| JIEJJGMNDWIGBJ-UHFFFAOYSA-N | $6.2\times10^{-3}$ | | Raventos-Duran et al. (2010) | Q | 244 |
| | $3.9\times10^{-3}$ | | Raventos-Duran et al. (2010) | Q | 245 |
| | $5.3\times10^{-3}$ | | Gharagheizi et al. (2010) | Q | 246 |
| | $6.3\times10^{-3}$ | | Modarresi et al. (2007) | Q | 67 |
| | $1.2\times10^{-2}$ | | Katritzky et al. (1998) | Q | |
| 2-ethoxy-2-methylbutane | $5.0\times10^{-3}$ | 7400 | Brockbank (2013) | L | 1 |
| $C_7H_{16}O$ | $4.8\times10^{-3}$ | 7600 | Haimi et al. (2006) | M | 525 |
| (*tert*-amyl ethyl ether) | $5.2\times10^{-4}$ | | Duchowicz et al. (2020) | V | 186 |
| [919-94-8] | $1.2\times10^{-3}$ | | Duchowicz et al. (2020) | Q | |
| KFRVYYGHSPLXSZ-UHFFFAOYSA-N | $2.5\times10^{-3}$ | | Raventos-Duran et al. (2010) | Q | 242, 243 |
| | $3.1\times10^{-3}$ | | Raventos-Duran et al. (2010) | Q | 244 |
| | $2.5\times10^{-3}$ | | Raventos-Duran et al. (2010) | Q | 245 |
| | $2.4\times10^{-3}$ | | Hilal et al. (2008) | Q | |
| | $1.1\times10^{-2}$ | | Modarresi et al. (2007) | Q | 67 |
| *tert*-butyl isopropyl ether | $6.6\times10^{-4}$ | | Yaws (1999) | ? | 21 |
| $C_7H_{16}O$ | | | | | |
| [17348-59-3] | | | | | |
| HNFSPSWQNZVCTB-UHFFFAOYSA-N | | | | | |
| dipropylene glycol monomethyl ether | $8.6\times10^{3}$ | | Bartelt-Hunt et al. (2008) | ? | 21 |
| $C_7H_{16}O_3$ | | | | | |
| [34590-94-8] | | | | | |
| WGYZMNBUZFHYRX-UHFFFAOYSA-N | | | | | |
| dibutyl ether | $2.1\times10^{-3}$ | 6700 | Brockbank (2013) | L | 1 |
| $C_4H_9OC_4H_9$ | $7.2\times10^{-3}$ | 10000 | Hwang et al. (2010) | M | 519, 11 |
| [142-96-1] | $2.2\times10^{-3}$ | | Li and Carr (1993) | M | |
| DURPTKYDGMDSBL-UHFFFAOYSA-N | $1.3\times10^{-3}$ | | Li et al. (1993) | M | |
| | $2.1\times10^{-3}$ | | Ioffe et al. (1984) | M | |
| | $2.1\times10^{-3}$ | | Mackay et al. (2006c) | V | |
| | $2.1\times10^{-3}$ | | Mackay et al. (1993) | V | |
| | $1.6\times10^{-3}$ | | Pierotti et al. (1959) | X | 414 |
| | $1.4\times10^{-2}$ | | Keshavarz et al. (2022) | Q | |
| | $7.4\times10^{-3}$ | | Duchowicz et al. (2020) | Q | |
| | $1.6\times10^{-3}$ | | Li et al. (2014) | Q | 241 |
| | $2.8\times10^{-3}$ | | Gharagheizi et al. (2012) | Q | |
| | $2.0\times10^{-3}$ | | Raventos-Duran et al. (2010) | Q | 242, 243 |
| | $6.2\times10^{-3}$ | | Raventos-Duran et al. (2010) | Q | 244 |
| | $2.0\times10^{-3}$ | | Raventos-Duran et al. (2010) | Q | 245 |
| | $3.1\times10^{-3}$ | | Hilal et al. (2008) | Q | |
| | $6.1\times10^{-3}$ | | Modarresi et al. (2007) | Q | 67 |
| | | 6600 | Kühne et al. (2005) | Q | |
| | $1.8\times10^{-3}$ | | Yaffe et al. (2003) | Q | 248, 249 |



Table A3.9: Ethers (ROR) (...continued)

| Substance Formula (Trivial Name) [CAS Registry Number] InChIKey | $H_s^{cp}$ (at $T^{\ominus}$) $\left[\dfrac{\text{mol}}{\text{m}^3\,\text{Pa}}\right]$ | $\dfrac{\text{d}\ln H_s^{cp}}{\text{d}(1/T)}$ [K] | Reference | Type | Note |
|---|---|---|---|---|---|
| | $9.9\times10^{-3}$ | | Katritzky et al. (1998) | Q | |
| | $6.4\times10^{-4}$ | | Nirmalakhandan et al. (1997) | Q | |
| | $2.2\times10^{-3}$ | | Russell et al. (1992) | Q | 279 |
| | $2.5\times10^{-3}$ | | Suzuki et al. (1992) | Q | 232 |
| | $1.6\times10^{-3}$ | | Duchowicz et al. (2020) | ? | 185, 21 |
| | | 7000 | Kühne et al. (2005) | ? | |
| | $1.8\times10^{-2}$ | | Yaws (1999) | ? | 21 |
| | $1.6\times10^{-3}$ | | Abraham et al. (1990) | ? | |
| diisobutyl ether $C_8H_{18}O$ [628-55-7] SZNYYWIUQFZLLT-UHFFFAOYSA-N | $3.7\times10^{-3}$ | | Hilal et al. (2008) | Q | |
| di-*sec*-butyl ether $C_8H_{18}O$ [6863-58-7] HHBZZTKMMLDNDN-UHFFFAOYSA-N | $1.8\times10^{-3}$ $1.8\times10^{-3}$ $1.0\times10^{-3}$ $8.0\times10^{-3}$ $1.7\times10^{-3}$ $2.8\times10^{-3}$ | | Yaws (2003) Yaws (2003) Dupeux et al. (2022) Gharagheizi et al. (2012) Gharagheizi et al. (2010) Yaws (1999) | X X Q Q Q ? | 258 237 259 246 21 |
| di-*tert*-butyl ether $C_8H_{18}O$ [6163-66-2] AQEFLFZSWDEAIP-UHFFFAOYSA-N | $2.1\times10^{-3}$ $1.1\times10^{-3}$ $3.0\times10^{-3}$ | | Yaws (2003) Gharagheizi et al. (2010) Yaws (1999) | X Q ? | 237 246 21 |
| 2-methoxy-2,4,4-trimethylpentane $C_9H_{20}O$ (methyl *tert*-octyl ether) [62108-41-2] IKZVAPMTXDXWMX-UHFFFAOYSA-N | $1.9\times10^{-3}$ | | Ebert et al. (2023) | ? | 526 |
| 2-methoxy-2-methylheptane $C_9H_{20}O$ [76589-16-7] KJRACWZCOVHBQU-UHFFFAOYSA-N | $1.9\times10^{-3}$ | 8100 | Haimi et al. (2006) | M | 527 |
| 2-ethoxy-2,4,4-trimethylpentane $C_{10}H_{22}O$ (ethyl tert-octyl ether) [187103-12-4] JGPJRBWLBUGAQN-UHFFFAOYSA-N | $7.7\times10^{-4}$ | 6900 | Haimi et al. (2006) | M | 528 |
| 1,1'-oxybispentane $C_{10}H_{22}O$ (dipentyl ether) [693-65-2] AOPDRZXCEAKHHW-UHFFFAOYSA-N | $3.5\times10^{-4}$ $3.5\times10^{-4}$ $4.5\times10^{-4}$ $4.3\times10^{-3}$ $4.3\times10^{-4}$ $2.7\times10^{-3}$ $1.0\times10^{-3}$ | | Yaws (2003) Yaws (2003) Dupeux et al. (2022) Gharagheizi et al. (2012) Gharagheizi et al. (2010) Hilal et al. (2008) Yaws (1999) Brockbank (2013) | X X Q Q Q Q ? W | 258 237 259 246 21 529 |





Table A3.9: Ethers (ROR) (...continued)

| Substance<br>Formula<br>(Trivial Name)<br>[CAS Registry Number]<br>InChIKey | $H_s^{cp}$<br>(at $T^\ominus$)<br>$\left[\dfrac{\mathrm{mol}}{\mathrm{m^3\,Pa}}\right]$ | $\dfrac{\mathrm{d}\ln H_s^{cp}}{\mathrm{d}(1/T)}$<br><br>[K] | Reference | Type | Note |
|---|---|---|---|---|---|
| 1,1'-oxybis(3-methylbutane) | $6.8\times10^{-3}$ | | Duchowicz et al. (2020) | V | 186 |
| $C_{10}H_{22}O$ | $6.6\times10^{-3}$ | | HSDB (2015) | V | |
| (diisopentyl ether) | $1.1\times10^{-3}$ | | Duchowicz et al. (2020) | Q | |
| [544-01-4] | $1.2\times10^{-3}$ | | Raventos-Duran et al. (2010) | Q | 242, 243 |
| AQZGPSLYZOOYQP-UHFFFAOYSA-N | $4.9\times10^{-3}$ | | Raventos-Duran et al. (2010) | Q | 244 |
| | $1.2\times10^{-3}$ | | Raventos-Duran et al. (2010) | Q | 245 |
| | $3.3\times10^{-3}$ | | Hilal et al. (2008) | Q | |
| | $4.8\times10^{-3}$ | | Modarresi et al. (2007) | Q | 67 |
| 1,1'-oxybishexane | $2.9\times10^{-4}$ | | Yaws (2003) | X | 237 |
| $C_{12}H_{26}O$ | $5.6\times10^{-3}$ | | Gharagheizi et al. (2012) | Q | |
| (dihexyl ether) | $1.7\times10^{-4}$ | | Gharagheizi et al. (2010) | Q | 246 |
| [112-58-3] | $1.8\times10^{-3}$ | | Hilal et al. (2008) | Q | |
| BPIUIOXAFBGMNB-UHFFFAOYSA-N | $3.0\times10^{-3}$ | | Yaws (1999) | ? | 21 |
| | | | Brockbank (2013) | W | 530 |
| 1-ethoxy-3,7-dimethyloctane | $6.7\times10^{-4}$ | | Zhang et al. (2010) | Q | 287, 288 |
| $C_{12}H_{26}O$ | $5.3\times10^{-3}$ | | Zhang et al. (2010) | Q | 287, 289 |
| [22810-10-2] | $1.3\times10^{-3}$ | | Zhang et al. (2010) | Q | 287, 290 |
| HCHHIPCZJSRFRT-UHFFFAOYSA-N | $2.5\times10^{-4}$ | | Zhang et al. (2010) | Q | 287, 291 |
| diheptyl ether | $2.8\times10^{-5}$ | | Yaws (2003) | X | 237 |
| $C_{14}H_{30}O$ | $8.5\times10^{-5}$ | | Gharagheizi et al. (2010) | Q | 246 |
| [629-64-1] | | | | | |
| UJEGHEMJVNQWOJ-UHFFFAOYSA-N | | | | | |
| dioctyl ether | $1.3\times10^{-4}$ | | Yaws (2003) | X | 237 |
| $C_{16}H_{34}O$ | $6.7\times10^{-5}$ | | Gharagheizi et al. (2010) | Q | 246 |
| [629-82-3] | $8.2\times10^{-2}$ | | Yaws (1999) | ? | 21 |
| NKJOXAZJBOMXID-UHFFFAOYSA-N | | | | | |
| methoxycyclohexane | $3.1\times10^{-2}$ | | Hilal et al. (2008) | Q | |
| $C_7H_{14}O$ | | | | | |
| [931-56-6] | | | | | |
| GHDIHPNJQVDFBL-UHFFFAOYSA-N | | | | | |
| methyl cedryl ether | $2.5\times10^{-3}$ | | Zhang et al. (2010) | Q | 287, 288 |
| $C_{16}H_{28}O$ | $2.4\times10^{-3}$ | | Zhang et al. (2010) | Q | 287, 289 |
| [19870-74-7] | $7.7\times10^{-3}$ | | Zhang et al. (2010) | Q | 287, 290 |
| HRGPYCVTDOECMG-WALBABNVSA-N | $1.2\times10^{-3}$ | | Zhang et al. (2010) | Q | 287, 291 |
| dimethoxymethane | $6.1\times10^{-2}$ | 4700 | Brockbank (2013) | L | 1 |
| $CH_3OCH_2OCH_3$ | $6.3\times10^{-2}$ | 4800 | Ondo and Dohnal (2007) | M | 1 |
| (methylal) | $6.1\times10^{-2}$ | | HSDB (2015) | V | |
| [109-87-5] | $5.7\times10^{-2}$ | | Pierotti et al. (1959) | X | 414 |
| NKDDWNXOKDWJAK-UHFFFAOYSA-N | $1.6\times10^{-2}$ | | Keshavarz et al. (2022) | Q | |
| | $4.6\times10^{-1}$ | | Duchowicz et al. (2020) | Q | 184 |
| | $7.1\times10^{-2}$ | | Wang et al. (2017) | Q | 80, 238 |
| | $3.6\times10^{-1}$ | | Wang et al. (2017) | Q | 80, 239 |
| | $6.5\times10^{-2}$ | | Wang et al. (2017) | Q | 80, 240 |



Table A3.9: Ethers (ROR) (... continued)

| Substance Formula (Trivial Name) [CAS Registry Number] InChIKey | $H_s^{cp}$ (at $T^\ominus$) $\left[\dfrac{\text{mol}}{\text{m}^3\,\text{Pa}}\right]$ | $\dfrac{\text{d}\ln H_s^{cp}}{\text{d}(1/T)}$ [K] | Reference | Type | Note |
|---|---|---|---|---|---|
|  | $1.2\times10^{-1}$ |  | Raventos-Duran et al. (2010) | Q | 242, 243 |
|  | $1.6\times10^{-1}$ |  | Raventos-Duran et al. (2010) | Q | 244 |
|  | $2.0\times10^{-1}$ |  | Raventos-Duran et al. (2010) | Q | 245 |
|  | $2.3\times10^{-1}$ |  | Hilal et al. (2008) | Q |  |
|  | $6.9\times10^{-2}$ |  | Modarresi et al. (2007) | Q | 67 |
|  | $5.7\times10^{-2}$ |  | Duchowicz et al. (2020) | ? | 185, 21 |
|  | $6.9\times10^{-2}$ |  | Yaws (1999) | ? | 21, 12 |
| trimethoxymethane HC(OCH$_3$)$_3$ [149-73-5] PYOKUURKVVELLB-UHFFFAOYSA-N | $6.9\times10^{-1}$ |  | Guthrie (1973) | V |  |
| diethoxymethane C$_5$H$_{12}$O$_2$ [462-95-3] KLKFAASOGCDTDT-UHFFFAOYSA-N | $1.5\times10^{-1}$ |  | Duchowicz et al. (2020) | V | 186 |
|  | $7.5\times10^{-2}$ |  | Duchowicz et al. (2020) | Q |  |
|  | $7.8\times10^{-2}$ |  | Raventos-Duran et al. (2010) | Q | 242, 243 |
|  | $1.6\times10^{-1}$ |  | Raventos-Duran et al. (2010) | Q | 244 |
|  | $9.9\times10^{-2}$ |  | Raventos-Duran et al. (2010) | Q | 245 |
|  | $7.2\times10^{-2}$ |  | Modarresi et al. (2007) | Q | 67 |
| 1,1-diethoxyethane (C$_2$H$_5$O)$_2$CHCH$_3$ (acetal) [105-57-7] DHKHKXVYLBGOIT-UHFFFAOYSA-N | $1.0\times10^{-1}$ |  | Duchowicz et al. (2020) | V | 186 |
|  | $1.0\times10^{-1}$ |  | HSDB (2015) | V |  |
|  | $1.0\times10^{-1}$ |  | Hine and Mookerjee (1975) | V |  |
|  | $1.2\times10^{-1}$ |  | Abney (2021) | Q | 399 |
|  | $3.2\times10^{-2}$ |  | Duchowicz et al. (2020) | Q |  |
|  | $5.7\times10^{-2}$ |  | Hilal et al. (2008) | Q |  |
|  | $6.9\times10^{-2}$ |  | Modarresi et al. (2007) | Q | 67 |
|  | $1.1\times10^{-1}$ |  | Yaws (1999) | ? | 21 |
| 1,2-diethoxyethane C$_2$H$_5$OC$_2$H$_4$OC$_2$H$_5$ [629-14-1] LZDKZFUFMNSQCJ-UHFFFAOYSA-N | $1.6\times10^{-1}$ |  | Duchowicz et al. (2020) | V | 186 |
|  | $1.6\times10^{-1}$ |  | HSDB (2015) | V |  |
|  | $1.6\times10^{-1}$ |  | Hine and Mookerjee (1975) | V |  |
|  | $1.2\times10^{-1}$ |  | Howard and Meylan (1997) | X | 446 |
|  | $8.3\times10^{-2}$ |  | Duchowicz et al. (2020) | Q |  |
|  | $3.9\times10^{-1}$ |  | Hilal et al. (2008) | Q |  |
|  | $2.1\times10^{-1}$ |  | Modarresi et al. (2007) | Q | 67 |
| 1,1,1-trimethoxyethane CH$_3$C(OCH$_3$)$_3$ [1445-45-0] HDPNBNXLBDFELL-UHFFFAOYSA-N | $6.4\times10^{-1}$ |  | Guthrie (1973) | V |  |
| 1,2-dimethoxyethane C$_4$H$_{10}$O$_2$ [110-71-4] XTHFKEDIFFGKHM-UHFFFAOYSA-N | 1.3 | 7300 | Brockbank (2013) | L | 1, 531 |
|  | 2.1 |  | O'Farrell and Waghorne (2010) | M |  |
|  | 1.4 | 7200 | Ondo and Dohnal (2007) | M | 1 |
|  | 1.4 | 7100 | Cabani et al. (1978) | T |  |
|  | 9.0 |  | HSDB (2015) | Q | 99 |
|  | $5.3\times10^{-1}$ |  | Hilal et al. (2008) | Q |  |



Table A3.9: Ethers (ROR) (...continued)

| Substance Formula (Trivial Name) [CAS Registry Number] InChIKey | $H_s^{cp}$ (at $T^{\ominus}$) $\left[\dfrac{\text{mol}}{\text{m}^3\,\text{Pa}}\right]$ | $\dfrac{\mathrm{d}\ln H_s^{cp}}{\mathrm{d}(1/T)}$ [K] | Reference | Type | Note |
|---|---|---|---|---|---|
| 3-oxa-1-hexanol | $2.0\times10^1$ | 8400 | Cabani et al. (1978) | T | |
| $C_5H_{12}O_2$ | $6.6\times10^2$ | | HSDB (2015) | Q | 99 |
| (2-propoxyethanol) | $2.0\times10^1$ | | Raventos-Duran et al. (2010) | Q | 242, 243 |
| [2807-30-9] | 9.9 | | Raventos-Duran et al. (2010) | Q | 244 |
| YEYKMVJDLWJFOA-UHFFFAOYSA-N | $1.2\times10^2$ | | Raventos-Duran et al. (2010) | Q | 245 |
| | $1.0\times10^1$ | | Hilal et al. (2008) | Q | |
| | 5.8 | | Nirmalakhandan et al. (1997) | Q | |
| 3-oxa-1-heptanol | $1.3\times10^1$ | 8300 | Brockbank (2013) | L | 1 |
| $C_6H_{14}O_2$ | 3.5 | 7700 | Hiatt (2013) | M | |
| (2-butoxyethanol; butyl cellosolve) | $1.3\times10^1$ | 8300 | Kim et al. (2000) | M | 33 |
| [111-76-2] | 2.7 | | Johanson and Dynésius (1988) | M | 14 |
| POAOYUHQDCAZBD-UHFFFAOYSA-N | $1.6\times10^1$ | 8900 | Cabani et al. (1978) | T | |
| | $1.7\times10^1$ | | Keshavarz et al. (2022) | Q | |
| | 7.4 | | Duchowicz et al. (2020) | Q | 184 |
| | 5.4 | | Wang et al. (2017) | Q | 80, 238 |
| | $1.9\times10^1$ | | Wang et al. (2017) | Q | 80, 239 |
| | 3.2 | | Wang et al. (2017) | Q | 80, 240 |
| | $1.6\times10^1$ | | Raventos-Duran et al. (2010) | Q | 271, 243 |
| | 7.8 | | Raventos-Duran et al. (2010) | Q | 244 |
| | $9.9\times10^1$ | | Raventos-Duran et al. (2010) | Q | 245 |
| | 7.7 | | Hilal et al. (2008) | Q | |
| | $1.9\times10^1$ | | Modarresi et al. (2007) | Q | 67 |
| | 4.5 | | Nirmalakhandan et al. (1997) | Q | |
| | 6.2 | | Duchowicz et al. (2020) | ? | 185, 21 |
| | $2.8\times10^1$ | | Yaws (1999) | ? | 21, 12 |
| 1-methoxy-2-propanol | 4.8 | | Johanson and Dynésius (1988) | M | 14 |
| $C_4H_{10}O_2$ | $1.1\times10^1$ | | Hilal et al. (2008) | C | |
| [107-98-2] | 9.2 | | Keshavarz et al. (2022) | Q | |
| ARXJGSRGQADJSQ-UHFFFAOYSA-N | 6.7 | | Duchowicz et al. (2020) | Q | 299 |
| | 8.3 | | Wang et al. (2017) | Q | 80, 238 |
| | $4.7\times10^1$ | | Wang et al. (2017) | Q | 80, 239 |
| | 4.9 | | Wang et al. (2017) | Q | 80, 240 |
| | $2.5\times10^1$ | | Raventos-Duran et al. (2010) | Q | 242, 243 |
| | $2.0\times10^1$ | | Raventos-Duran et al. (2010) | Q | 244 |
| | $1.6\times10^2$ | | Raventos-Duran et al. (2010) | Q | 245 |
| | $1.2\times10^1$ | | Hilal et al. (2008) | Q | |
| | $2.8\times10^1$ | | Modarresi et al. (2007) | Q | 67 |
| | $1.1\times10^1$ | | Yaffe et al. (2003) | Q | 248, 249 |
| | 3.5 | | Katritzky et al. (1998) | Q | |
| | $1.1\times10^1$ | | Duchowicz et al. (2020) | ? | 185, 21 |



Table A3.9: Ethers (ROR) (. . . continued)

| Substance Formula (Trivial Name) [CAS Registry Number] InChIKey | $H_s^{cp}$ (at $T^{\ominus}$) $\left[\dfrac{\text{mol}}{\text{m}^3\,\text{Pa}}\right]$ | $\dfrac{\mathrm{d}\ln H_s^{cp}}{\mathrm{d}(1/T)}$ [K] | Reference | Type | Note |
|---|---|---|---|---|---|
| 4-methyl-3-oxa-1-pentanol $C_5H_{12}O_2$ (2-isopropoxyethanol) [109-59-1] HCGFUIQPSOCUHI-UHFFFAOYSA-N | 4.8 $1.2\times10^1$ 2.6 $2.0\times10^1$ 7.8 $1.2\times10^2$ 7.9 $2.7\times10^1$ $1.1\times10^1$ | | Johanson and Dynésius (1988) Keshavarz et al. (2022) Duchowicz et al. (2020) Raventos-Duran et al. (2010) Raventos-Duran et al. (2010) Raventos-Duran et al. (2010) Hilal et al. (2008) Modarresi et al. (2007) Duchowicz et al. (2020) | M Q Q Q Q Q Q Q ? | 14 184 271, 243 244 245 67 185, 21 |
| 1,2-dibutoxyethane $C_{10}H_{22}O_2$ [112-48-1] GDXHBFHOEYVPED-UHFFFAOYSA-N | $9.9\times10^{-1}$ $1.4\times10^{-1}$ $1.1\times10^{-1}$ | | HSDB (2015) Hilal et al. (2008) Modarresi et al. (2007) | V Q Q | 67 |
| 3,6-dioxa-1-decanol $C_8H_{18}O_3$ (butyl carbitol) [112-34-5] OAYXUHPQHDHDDZ-UHFFFAOYSA-N | $1.4\times10^3$ | | Kim et al. (2000) | M | |
| diethylene glycol dibutyl ether $C_{12}H_{26}O_3$ [112-73-2] KZVBBTZJMSWGTK-UHFFFAOYSA-N | 2.5 3.9 6.1 1.3 6.0 3.5 1.5 4.0 | 13000 | Brockbank (2013) Duchowicz et al. (2020) Yaws (2003) Duchowicz et al. (2020) Gharagheizi et al. (2010) Hilal et al. (2008) Modarresi et al. (2007) Yaws (1999) | L V X Q Q Q Q ? | 1 186 237, 12 246 67 21, 12 |
| methyl vinyl ether $C_3H_6O$ [107-25-5] XJRBAMWJDBPFIM-UHFFFAOYSA-N | $1.5\times10^{-3}$ $9.9\times10^{-3}$ $6.2\times10^{-4}$ $1.6\times10^{-3}$ | | HSDB (2015) Raventos-Duran et al. (2010) Raventos-Duran et al. (2010) Raventos-Duran et al. (2010) | V Q Q Q | 242, 243 244 245 |
| ethyl vinyl ether $C_4H_8O$ [109-92-2] FJKIXWOMBXYWOQ-UHFFFAOYSA-N | $1.8\times10^{-3}$ $2.3\times10^{-2}$ $6.2\times10^{-3}$ $3.9\times10^{-4}$ $1.2\times10^{-3}$ | | Duchowicz et al. (2020) Duchowicz et al. (2020) Raventos-Duran et al. (2010) Raventos-Duran et al. (2010) Raventos-Duran et al. (2010) | V Q Q Q Q | 186 242, 243 244 245 |
| propyl vinyl ether $C_5H_{10}O$ [764-47-6] OVGRCEFMXPHEBL-UHFFFAOYSA-N | $3.1\times10^{-3}$ | 4800 | Hwang et al. (2010) | M | 519, 11 |
| allyl glycidyl ether $C_6H_{10}O_2$ [106-92-3] LSWYGACWGAICNM-UHFFFAOYSA-N | 2.1 | | Ebert et al. (2023) | ? | 316 |





Table A3.9: Ethers (ROR) (...continued)

| Substance Formula (Trivial Name) [CAS Registry Number] InChIKey | $H_s^{cp}$ (at $T^{\ominus}$) $\left[\dfrac{\mathrm{mol}}{\mathrm{m}^3\,\mathrm{Pa}}\right]$ | $\dfrac{\mathrm{d}\ln H_s^{cp}}{\mathrm{d}(1/T)}$ [K] | Reference | Type | Note |
|---|---|---|---|---|---|
| butyl vinyl ether | $4.6\times10^{-3}$ | | Duchowicz et al. (2020) | V | 186 |
| $C_6H_{12}O$ | $4.5\times10^{-3}$ | | HSDB (2015) | V | |
| [111-34-2] | $7.3\times10^{-4}$ | | Dupeux et al. (2022) | Q | 259 |
| UZKWTJUDCOPSNM-UHFFFAOYSA-N | $2.5\times10^{-2}$ | | Duchowicz et al. (2020) | Q | |
| | $3.9\times10^{-3}$ | | Raventos-Duran et al. (2010) | Q | 242, 243 |
| | $2.0\times10^{-4}$ | | Raventos-Duran et al. (2010) | Q | 244 |
| | $7.8\times10^{-4}$ | | Raventos-Duran et al. (2010) | Q | 245 |
| | $2.5\times10^{-3}$ | | Modarresi et al. (2007) | Q | 67 |
| isobutyl vinyl ether | $3.9\times10^{-3}$ | | Raventos-Duran et al. (2010) | Q | 242, 243 |
| $C_6H_{12}O$ | $2.5\times10^{-4}$ | | Raventos-Duran et al. (2010) | Q | 244 |
| [109-53-5] | $7.8\times10^{-4}$ | | Raventos-Duran et al. (2010) | Q | 245 |
| OZCMOJQQLBXBKI-UHFFFAOYSA-N | $1.9\times10^{-4}$ | | Hilal et al. (2008) | Q | |
| | $2.5\times10^{-3}$ | | Modarresi et al. (2007) | Q | 67 |
| benzyl methyl ether | $1.1\times10^{-1}$ | | Duchowicz et al. (2020) | V | 186 |
| $C_8H_{10}O$ | $1.3\times10^{-1}$ | | Duchowicz et al. (2020) | Q | |
| [538-86-3] | | | | | |
| GQKZBCPTCWJTAS-UHFFFAOYSA-N | | | | | |
| benzyl ethyl ether | $6.7\times10^{-2}$ | | Dupeux et al. (2022) | Q | 259 |
| $C_9H_{12}O$ | | | | | |
| [539-30-0] | | | | | |
| AXPZDYVDTMMLNB-UHFFFAOYSA-N | | | | | |
| methoxybenzene | $2.9\times10^{-2}$ | 4200 | Brockbank (2013) | L | 1 |
| $C_6H_5OCH_3$ | $2.9\times10^{-2}$ | 4200 | Brockbank et al. (2013) | M | |
| (anisole) | $2.6\times10^{-2}$ | 4800 | Dewulf et al. (1999) | M | |
| [100-66-3] | $3.2\times10^{-2}$ | | Li and Carr (1993) | M | |
| RDOXTESZEPMUJZ-UHFFFAOYSA-N | $2.0\times10^{-2}$ | | Duchowicz et al. (2020) | V | 186 |
| | $3.1\times10^{-2}$ | | Mackay et al. (2006c) | V | |
| | $4.0\times10^{-2}$ | | Mackay et al. (1993) | V | |
| | $2.3\times10^{-3}$ | | Hine and Mookerjee (1975) | V | |
| | $2.3\times10^{-3}$ | | Hine and Weimar (1965) | R | |
| | $2.7\times10^{-3}$ | | Yaws (2003) | X | 258 |
| | $2.7\times10^{-3}$ | | Yaws (2003) | X | 237 |
| | $6.9\times10^{-2}$ | | Schüürmann (2000) | C | 21 |
| | $1.9\times10^{-2}$ | | Dupeux et al. (2022) | Q | 259 |
| | $1.2\times10^{-1}$ | | Duchowicz et al. (2020) | Q | |
| | $2.3\times10^{-3}$ | | HSDB (2015) | Q | 99 |
| | $1.9\times10^{-2}$ | | Li et al. (2014) | Q | 241 |
| | $1.3\times10^{-1}$ | | Gharagheizi et al. (2010) | Q | 246 |
| | $9.0\times10^{-3}$ | | Hilal et al. (2008) | Q | |
| | $3.3\times10^{-2}$ | | Modarresi et al. (2007) | Q | 67 |
| | | 4500 | Kühne et al. (2005) | Q | |
| | $8.6\times10^{-3}$ | | Yao et al. (2002) | Q | 229 |
| | $1.2\times10^{-2}$ | | Nirmalakhandan et al. (1997) | Q | |
| | $2.3\times10^{-3}$ | | Suzuki et al. (1992) | Q | 232 |
| | | 4300 | Kühne et al. (2005) | ? | |



Table A3.9: Ethers (ROR) (...continued)

| Substance Formula (Trivial Name) [CAS Registry Number] InChIKey | $H_s^{cp}$ (at $T^{\ominus}$) $\left[\dfrac{\mathrm{mol}}{\mathrm{m^3\,Pa}}\right]$ | $\dfrac{\mathrm{d}\ln H_s^{cp}}{\mathrm{d}(1/T)}$ [K] | Reference | Type | Note |
|---|---|---|---|---|---|
| | $2.7\times10^{-3}$ | | Yaws (1999) | ? | 21 |
| | $2.5\times10^{-2}$ | | Abraham et al. (1990) | ? | |
| ethoxybenzene | $1.7\times10^{-2}$ | | Li and Carr (1993) | M | |
| $C_8H_{10}O$ | $2.2\times10^{-2}$ | | Duchowicz et al. (2020) | V | 186 |
| (phenetole) | $2.2\times10^{-2}$ | | HSDB (2015) | V | |
| [103-73-1] | $2.3\times10^{-2}$ | | Mackay et al. (2006c) | V | |
| DLRJIFUOBPOJNS-UHFFFAOYSA-N | $4.4\times10^{-2}$ | | Duchowicz et al. (2020) | Q | |
| | $4.9\times10^{-2}$ | | Raventos-Duran et al. (2010) | Q | 242, 243 |
| | $7.8\times10^{-3}$ | | Raventos-Duran et al. (2010) | Q | 244 |
| | $2.5\times10^{-2}$ | | Raventos-Duran et al. (2010) | Q | 245 |
| | $6.5\times10^{-3}$ | | Hilal et al. (2008) | Q | |
| | $2.9\times10^{-2}$ | | Modarresi et al. (2007) | Q | 67 |
| | $2.6\times10^{-1}$ | | Katritzky et al. (1998) | Q | |
| | $1.0\times10^{-2}$ | | Nirmalakhandan et al. (1997) | Q | |
| | $1.7\times10^{-2}$ | | Abraham et al. (1990) | ? | |
| 1,2-dimethoxybenzene | | 5100 | Kühne et al. (2005) | Q | |
| $C_8H_{10}O_2$ | | 2400 | Kühne et al. (2005) | ? | |
| [91-16-7] | | | | | |
| ABDKAPXRBAPSQN-UHFFFAOYSA-N | | | | | |
| 2-phenoxyethanol | $2.1\times10^2$ | | Duchowicz et al. (2020) | V | 186 |
| $C_8H_{10}O_2$ | $2.0\times10^2$ | | HSDB (2015) | V | |
| [122-99-6] | $4.0\times10^1$ | | Duchowicz et al. (2020) | Q | |
| QCDWFXQBSFUVSP-UHFFFAOYSA-N | $3.4\times10^1$ | | Hilal et al. (2008) | Q | |
| | $1.4\times10^2$ | | Modarresi et al. (2007) | Q | 67 |
| 2-(phenylmethoxy)-ethanol | $8.2\times10^1$ | | Duchowicz et al. (2020) | V | 186 |
| $C_9H_{12}O_2$ | $4.4\times10^1$ | | Duchowicz et al. (2020) | Q | |
| [622-08-2] | $1.6\times10^3$ | | Raventos-Duran et al. (2010) | Q | 242, 243 |
| CUZKCNWZBXLAJX-UHFFFAOYSA-N | $1.6\times10^2$ | | Raventos-Duran et al. (2010) | Q | 244 |
| | $3.1\times10^3$ | | Raventos-Duran et al. (2010) | Q | 245 |
| | $1.5\times10^2$ | | Hilal et al. (2008) | Q | |
| | $7.0\times10^2$ | | Modarresi et al. (2007) | Q | 67 |
| 1,2,3-trimethoxybenzene | 3.6 | | Schüürmann (2000) | V | |
| $C_9H_{12}O_3$ | | | | | |
| [634-36-6] | | | | | |
| CRUILBNAQILVHZ-UHFFFAOYSA-N | | | | | |
| 1-methoxy-4-(1-propenyl)-benzene | $9.9\times10^{-2}$ | 6200 | van Roon et al. (2005) | V | |
| $C_{10}H_{12}O$ | $1.0\times10^{-1}$ | | Yaws (2003) | X | 258 |
| (anethole) | $3.7\times10^{-2}$ | | Dupeux et al. (2022) | Q | 259 |
| [104-46-1] | $1.4\times10^{-1}$ | | HSDB (2015) | Q | 99 |
| RUVINXPYWBROJD-UHFFFAOYSA-N | $2.0\times10^{-2}$ | | Hilal et al. (2008) | Q | |
| | $1.0\times10^{-1}$ | | Yaws (1999) | ? | 21 |



Table A3.9: Ethers (ROR) (...continued)

| Substance Formula (Trivial Name) [CAS Registry Number] InChIKey | $H_s^{cp}$ (at $T^{\ominus}$) $\left[\dfrac{\text{mol}}{\text{m}^3\,\text{Pa}}\right]$ | $\dfrac{\text{d}\ln H_s^{cp}}{\text{d}(1/T)}$ [K] | Reference | Type | Note |
|---|---|---|---|---|---|
| 2-methoxy-4-(2-propenyl)-phenol | 9.2 | | McFall et al. (2020) | M | |
| $C_{10}H_{12}O_2$ | 5.0 | | Duchowicz et al. (2020) | V | 186 |
| (eugenol) | 6.9 | | Martins et al. (2017) | V | 315 |
| [97-53-0] | 5.1 | | HSDB (2015) | V | |
| RRAFCDWBNXTKKO-UHFFFAOYSA-N | 7.2 | 9700 | van Roon et al. (2005) | V | |
| | 3.9 | | Dupeux et al. (2022) | Q | 259 |
| | $3.5\times10^{-1}$ | | Abney (2021) | Q | 399 |
| | $2.0\times10^{1}$ | | Duchowicz et al. (2020) | Q | |
| | 5.7 | | McFall et al. (2020) | Q | 474 |
| 1,2-dimethoxy-4-(2-propenyl)-benzene | 1.8 | | Duchowicz et al. (2020) | V | 186 |
| $C_{11}H_{14}O_2$ | 1.8 | | HSDB (2015) | V | |
| [93-15-2] | 1.0 | | Duchowicz et al. (2020) | Q | |
| ZYEMGPIYFIJGTP-UHFFFAOYSA-N | 3.6 | | Hilal et al. (2008) | Q | |
| | $1.9\times10^{-1}$ | | Modarresi et al. (2007) | Q | 67 |
| diphenyl ether | $1.9\times10^{-2}$ | | Brockbank (2013) | L | |
| $C_{12}H_{10}O$ | $3.5\times10^{-2}$ | | Duchowicz et al. (2020) | V | 186 |
| [101-84-8] | $3.5\times10^{-2}$ | | HSDB (2015) | V | |
| USIUVYZYUHIAEV-UHFFFAOYSA-N | $3.7\times10^{-2}$ | | Mackay et al. (2006c) | V | |
| | $1.1\times10^{-1}$ | | Kurz and Ballschmiter (1999) | V | |
| | $3.7\times10^{-2}$ | | Mackay et al. (1993) | V | |
| | $1.8\times10^{-2}$ | | Yaws (2003) | X | 237 |
| | $2.9\times10^{-1}$ | | Duchowicz et al. (2020) | Q | |
| | $1.9\times10^{-2}$ | | Gharagheizi et al. (2010) | Q | 246 |
| | $1.7\times10^{-2}$ | | Hilal et al. (2008) | Q | |
| | $1.7\times10^{-1}$ | | Modarresi et al. (2007) | Q | 67 |
| | $1.9\times10^{-2}$ | | Yaws (1999) | ? | 21 |
| (phenoxymethyl)-oxirane | $1.2\times10^{1}$ | | Duchowicz et al. (2020) | V | 186 |
| $C_9H_{10}O_2$ | $1.2\times10^{1}$ | | HSDB (2015) | V | |
| [122-60-1] | 4.8 | | Duchowicz et al. (2020) | Q | |
| FQYUMYWMJTYZTK-UHFFFAOYSA-N | $6.1\times10^{-1}$ | | Hilal et al. (2008) | Q | |
| | 1.9 | | Modarresi et al. (2007) | Q | 67 |
| 1-dodecyl-4-phenoxybenzene | $3.4\times10^{-3}$ | | Zhang et al. (2010) | Q | 287, 288 |
| $C_{24}H_{34}O$ | $1.4\times10^{-3}$ | | Zhang et al. (2010) | Q | 287, 289 |
| [119345-02-7] | $1.7\times10^{-2}$ | | Zhang et al. (2010) | Q | 287, 290 |
| XSAHYEQPUFJGKW-UHFFFAOYSA-N | $7.7\times10^{-2}$ | | Zhang et al. (2010) | Q | 287, 291 |
| 2,2,4-trimethyl-4-(4-(4-(2,4,4-trimethylpentan-2-yl)phenoxy)phenyl)pentane | $1.3\times10^{-3}$ | | Zhang et al. (2010) | Q | 287, 288 |
| $C_{28}H_{42}O$ | $1.2\times10^{-3}$ | | Zhang et al. (2010) | Q | 287, 289 |
| [61702-88-3] | $5.4\times10^{-2}$ | | Zhang et al. (2010) | Q | 287, 290 |
| AJDONJVWDSZZQF-UHFFFAOYSA-N | $6.9\times10^{-3}$ | | Zhang et al. (2010) | Q | 287, 291 |



Table A3.9: Ethers (ROR) (...continued)

| Substance Formula (Trivial Name) [CAS Registry Number] InChIKey | $H_s^{cp}$ (at $T^{\ominus}$) $\left[\dfrac{\text{mol}}{\text{m}^3\,\text{Pa}}\right]$ | $\dfrac{\text{d}\ln H_s^{cp}}{\text{d}(1/T)}$ [K] | Reference | Type | Note |
|---|---|---|---|---|---|
| etofenprox C$_{25}$H$_{28}$O$_3$ [80844-07-1] YREQHYQNNWYQCJ-UHFFFAOYSA-N | $7.4\times10^1$ | | Maniere et al. (2011) | ? | 165 |
| di-*tert*-butyl *sec*-butylidene diperoxide | $1.2\times10^{-2}$ | | Zhang et al. (2010) | Q | 287, 288 |
| C$_{12}$H$_{26}$O$_4$ | $6.1\times10^{-5}$ | | Zhang et al. (2010) | Q | 287, 289 |
| [2167-23-9] | $1.6\times10^{-2}$ | | Zhang et al. (2010) | Q | 287, 290 |
| HQOVXPHOJANJBR-UHFFFAOYSA-N | 1.1 | | Zhang et al. (2010) | Q | 287, 291 |
| peroxide, 1,1-dimethylethyl 1-methyl-1-phenylethyl | $1.4\times10^{-2}$ | | Zhang et al. (2010) | Q | 287, 288 |
| C$_{13}$H$_{20}$O$_2$ | $4.8\times10^{-3}$ | | Zhang et al. (2010) | Q | 287, 289 |
| [3457-61-2] | $1.6\times10^{-2}$ | | Zhang et al. (2010) | Q | 287, 290 |
| BIISIZOQPWZPPS-UHFFFAOYSA-N | $1.9\times10^{-1}$ | | Zhang et al. (2010) | Q | 287, 291 |
| di-*tert*-butyl 1,1,4,4-tetramethyltetramethylene diperoxide | $3.9\times10^{-3}$ | | Zhang et al. (2010) | Q | 287, 288 |
| C$_{16}$H$_{34}$O$_4$ | $7.9\times10^{-4}$ | | Zhang et al. (2010) | Q | 287, 289 |
| [78-63-7] | $1.3\times10^{-1}$ | | Zhang et al. (2010) | Q | 287, 290 |
| DMWVYCCGCQPJEA-UHFFFAOYSA-N | $3.4\times10^{-1}$ | | Zhang et al. (2010) | Q | 287, 291 |
| 1,4-bis(1-*tert*-butylperoxy-1-methyl-ethyl)benzene | $1.0\times10^{-1}$ | | Zhang et al. (2010) | Q | 287, 288 |
| C$_{20}$H$_{34}$O$_4$ | $1.8\times10^{-2}$ | | Zhang et al. (2010) | Q | 287, 289 |
| [2781-00-2] | $2.9\times10^{-1}$ | | Zhang et al. (2010) | Q | 287, 290 |
| GWQOYRSARAWVTC-UHFFFAOYSA-N | 8.6 | | Zhang et al. (2010) | Q | 287, 291 |
| MCM:CH3OCH2OOH C$_2$H$_6$O$_3$ CDXAGPPPWKCPRI-UHFFFAOYSA-N | $2.2\times10^2$ $6.9\times10^1$ $7.1\times10^1$ | | Wang et al. (2017) Wang et al. (2017) Wang et al. (2017) | Q Q Q | 80, 238 80, 239 80, 240 |
| MCM:ETHOXOOH C$_2$H$_4$O$_3$ NOEKFNMWGHEGBN-UHFFFAOYSA-N | $7.1\times10^2$ $1.7\times10^2$ $1.6\times10^2$ | | Wang et al. (2017) Wang et al. (2017) Wang et al. (2017) | Q Q Q | 80, 238 80, 239 80, 240 |
| MCM:DMMAOOH C$_3$H$_8$O$_4$ SHLFZQKLTPTTSZ-UHFFFAOYSA-N | $5.1\times10^3$ $3.0\times10^2$ $1.2\times10^1$ | | Wang et al. (2017) Wang et al. (2017) Wang et al. (2017) | Q Q Q | 80, 238 80, 239 80, 240 |
| MCM:DMMBOOH C$_3$H$_8$O$_4$ HODDSHZVZWXLHR-UHFFFAOYSA-N | $5.5\times10^3$ $5.6\times10^1$ $2.2\times10^2$ | | Wang et al. (2017) Wang et al. (2017) Wang et al. (2017) | Q Q Q | 80, 238 80, 239 80, 240 |
| MCM:ETOMEOOH C$_3$H$_8$O$_3$ NZBMKAZNWPZJND-UHFFFAOYSA-N | $1.7\times10^2$ $3.9\times10^1$ $6.9\times10^1$ | | Wang et al. (2017) Wang et al. (2017) Wang et al. (2017) | Q Q Q | 80, 238 80, 239 80, 240 |





Table A3.9: Ethers (ROR) (. . . continued)

| Substance<br>Formula<br>(Trivial Name)<br>[CAS Registry Number]<br>InChIKey | $H_s^{cp}$<br>(at $T^{\ominus}$)<br>$\left[\dfrac{\mathrm{mol}}{\mathrm{m^3\,Pa}}\right]$ | $\dfrac{\mathrm{d}\ln H_s^{cp}}{\mathrm{d}(1/T)}$<br><br>[K] | Reference | Type | Note |
|---|---|---|---|---|---|
| MCM:MEMOXYCO3H | $2.6\times10^3$ | | Wang et al. (2017) | Q | 80, 238 |
| $C_3H_6O_4$ | $7.1\times10^1$ | | Wang et al. (2017) | Q | 80, 239 |
| PWWCDEMRYRBWOJ-UHFFFAOYSA-N | $1.1\times10^1$ | | Wang et al. (2017) | Q | 80, 240 |
| MCM:DIETETOOH | $1.7\times10^2$ | | Wang et al. (2017) | Q | 80, 238 |
| $C_4H_{10}O_3$ | $1.3\times10^1$ | | Wang et al. (2017) | Q | 80, 239 |
| CXWWPQGYBJCHJL-UHFFFAOYSA-N | $4.1\times10^1$ | | Wang et al. (2017) | Q | 80, 240 |
| MCM:ETOC2OOH | $1.6\times10^2$ | | Wang et al. (2017) | Q | 80, 238 |
| $C_4H_{10}O_3$ | $8.1\times10^1$ | | Wang et al. (2017) | Q | 80, 239 |
| NYVGIYPKZJSYSP-UHFFFAOYSA-N | $6.8\times10^1$ | | Wang et al. (2017) | Q | 80, 240 |
| MCM:ETOMECO3H | $2.1\times10^3$ | | Wang et al. (2017) | Q | 80, 238 |
| $C_4H_8O_4$ | $3.4\times10^1$ | | Wang et al. (2017) | Q | 80, 239 |
| ULYBJOMJQUHMOL-UHFFFAOYSA-N | $4.8$ | | Wang et al. (2017) | Q | 80, 240 |
| MCM:IPRMETOOH | $1.1\times10^2$ | | Wang et al. (2017) | Q | 80, 238 |
| $C_4H_{10}O_3$ | $4.5$ | | Wang et al. (2017) | Q | 80, 239 |
| PSBSELFEZHDPTA-UHFFFAOYSA-N | $2.8\times10^1$ | | Wang et al. (2017) | Q | 80, 240 |
| MCM:BOXMOOH | $1.3\times10^2$ | | Wang et al. (2017) | Q | 80, 238 |
| $C_5H_{12}O_3$ | $2.0\times10^1$ | | Wang et al. (2017) | Q | 80, 239 |
| DCHWALHTYQHXTG-UHFFFAOYSA-N | $1.6\times10^1$ | | Wang et al. (2017) | Q | 80, 240 |
| MCM:EIPEOOH | $8.9\times10^1$ | | Wang et al. (2017) | Q | 80, 238 |
| $C_5H_{12}O_3$ | $3.0$ | | Wang et al. (2017) | Q | 80, 239 |
| ZZXLAMSYBUVKJS-UHFFFAOYSA-N | $1.9\times10^1$ | | Wang et al. (2017) | Q | 80, 240 |
| MCM:IPROC21OOH | $1.7\times10^2$ | | Wang et al. (2017) | Q | 80, 238 |
| $C_5H_{12}O_3$ | $6.6$ | | Wang et al. (2017) | Q | 80, 239 |
| VGMUBGNCFVOQDX-UHFFFAOYSA-N | $2.4\times10^1$ | | Wang et al. (2017) | Q | 80, 240 |
| MCM:MTBEAOOH | $8.9\times10^1$ | | Wang et al. (2017) | Q | 80, 238 |
| $C_5H_{12}O_3$ | $8.5$ | | Wang et al. (2017) | Q | 80, 239 |
| BGZKVDIFKPQYHL-UHFFFAOYSA-N | $6.2\times10^1$ | | Wang et al. (2017) | Q | 80, 240 |
| MCM:MTBEBCO3H | $1.3\times10^3$ | | Wang et al. (2017) | Q | 80, 238 |
| $C_5H_{10}O_4$ | $4.7$ | | Wang et al. (2017) | Q | 80, 239 |
| NGYXBYFCPPVOQA-UHFFFAOYSA-N | $3.4\times10^{-1}$ | | Wang et al. (2017) | Q | 80, 240 |
| MCM:MTBEBOOH | $8.9\times10^1$ | | Wang et al. (2017) | Q | 80, 238 |
| $C_5H_{12}O_3$ | $2.5\times10^1$ | | Wang et al. (2017) | Q | 80, 239 |
| FPTBRWWJTBOWHN-UHFFFAOYSA-N | $2.0\times10^1$ | | Wang et al. (2017) | Q | 80, 240 |
| MCM:BOXMCO3H | $1.5\times10^3$ | | Wang et al. (2017) | Q | 80, 238 |
| $C_6H_{12}O_4$ | $1.7\times10^1$ | | Wang et al. (2017) | Q | 80, 239 |
| BMXLBBYCODRNIJ-UHFFFAOYSA-N | $8.3\times10^{-1}$ | | Wang et al. (2017) | Q | 80, 240 |
| MCM:DIIPRETOOH | $8.3\times10^1$ | | Wang et al. (2017) | Q | 80, 238 |
| $C_6H_{14}O_3$ | $1.6$ | | Wang et al. (2017) | Q | 80, 239 |
| PFVPTWUDXVSVAC-UHFFFAOYSA-N | $2.1\times10^1$ | | Wang et al. (2017) | Q | 80, 240 |



Table A3.9: Ethers (ROR) (...continued)

| Substance<br>Formula<br>(Trivial Name)<br>[CAS Registry Number]<br>InChIKey | $H_s^{cp}$<br>(at $T^{\ominus}$)<br>$\left[\dfrac{\text{mol}}{\text{m}^3\,\text{Pa}}\right]$ | $\dfrac{\text{d}\ln H_s^{cp}}{\text{d}(1/T)}$<br><br>[K] | Reference | Type | Note |
|---|---|---|---|---|---|
| MCM:ETBEACO3H<br>$C_6H_{12}O_4$<br>UAVAAIVEDVDYSQ-UHFFFAOYSA-N | $1.1\times10^3$<br>3.4<br>$3.1\times10^{-1}$ | | Wang et al. (2017)<br>Wang et al. (2017)<br>Wang et al. (2017) | Q<br>Q<br>Q | 80, 238<br>80, 239<br>80, 240 |
| MCM:ETBEAOOH<br>$C_6H_{14}O_3$<br>ITRUGVUCIYLYDK-UHFFFAOYSA-N | $8.3\times10^1$<br>$1.7\times10^1$<br>$1.7\times10^1$ | | Wang et al. (2017)<br>Wang et al. (2017)<br>Wang et al. (2017) | Q<br>Q<br>Q | 80, 238<br>80, 239<br>80, 240 |
| MCM:ETBEBOOH<br>$C_6H_{14}O_3$<br>VOZHKKMLSSYDIG-UHFFFAOYSA-N | $8.3\times10^1$<br>2.5<br>$2.8\times10^1$ | | Wang et al. (2017)<br>Wang et al. (2017)<br>Wang et al. (2017) | Q<br>Q<br>Q | 80, 238<br>80, 239<br>80, 240 |
| MCM:ETBECCO3H<br>$C_6H_{12}O_4$<br>AELSWALKLVZVPO-UHFFFAOYSA-N | $1.1\times10^3$<br>8.9<br>$1.1\times10^1$ | | Wang et al. (2017)<br>Wang et al. (2017)<br>Wang et al. (2017) | Q<br>Q<br>Q | 80, 238<br>80, 239<br>80, 240 |
| MCM:ETBECOOH<br>$C_6H_{14}O_3$<br>WJMPPRSRRBMJID-UHFFFAOYSA-N | $8.3\times10^1$<br>$2.0\times10^1$<br>$5.8\times10^1$ | | Wang et al. (2017)<br>Wang et al. (2017)<br>Wang et al. (2017) | Q<br>Q<br>Q | 80, 238<br>80, 239<br>80, 240 |
| MCM:IPROMC2OOH<br>$C_6H_{14}O_3$<br>JCIUPJDJCBRYHD-UHFFFAOYSA-N | $1.4\times10^2$<br>$2.2\times10^1$<br>$2.7\times10^1$ | | Wang et al. (2017)<br>Wang et al. (2017)<br>Wang et al. (2017) | Q<br>Q<br>Q | 80, 238<br>80, 239<br>80, 240 |
| MCM:IPROMCCO3H<br>$C_6H_{12}O_4$<br>SFJQIQZNOBLTBZ-UHFFFAOYSA-N | $1.8\times10^3$<br>7.8<br>$7.6\times10^{-1}$ | | Wang et al. (2017)<br>Wang et al. (2017)<br>Wang et al. (2017) | Q<br>Q<br>Q | 80, 238<br>80, 239<br>80, 240 |
| MCM:BCSOZ<br>$C_{15}H_{24}O_3$<br>DXZCAIRIBLPJLG-UHFFFAOYSA-N | 9.1<br>$2.0\times10^{-1}$<br>$7.4\times10^1$ | | Wang et al. (2017)<br>Wang et al. (2017)<br>Wang et al. (2017) | Q<br>Q<br>Q | 80, 238<br>80, 239<br>80, 240 |
| MCM:CH3OCH2OH<br>$C_2H_6O_2$<br>VHWYCFISAQVCCP-UHFFFAOYSA-N | 8.1<br>$5.4\times10^1$<br>$3.6\times10^1$ | | Wang et al. (2017)<br>Wang et al. (2017)<br>Wang et al. (2017) | Q<br>Q<br>Q | 80, 238<br>80, 239<br>80, 240 |
| MCM:DMMAOH<br>$C_3H_8O_3$<br>YANOHILBFNXRFM-UHFFFAOYSA-N | $1.7\times10^2$<br>$4.1\times10^2$<br>$1.2\times10^2$ | | Wang et al. (2017)<br>Wang et al. (2017)<br>Wang et al. (2017) | Q<br>Q<br>Q | 80, 238<br>80, 239<br>80, 240 |
| MCM:DMMBOH<br>$C_3H_8O_3$<br>IIGJYLXJNYBXEO-UHFFFAOYSA-N | $2.0\times10^2$<br>$4.6\times10^1$<br>$9.1\times10^1$ | | Wang et al. (2017)<br>Wang et al. (2017)<br>Wang et al. (2017) | Q<br>Q<br>Q | 80, 238<br>80, 239<br>80, 240 |
| MCM:ETOMEOH<br>$C_3H_8O_2$<br>RRLWYLINGKISHN-UHFFFAOYSA-N | 6.8<br>$3.0\times10^1$<br>$3.1\times10^1$ | | Wang et al. (2017)<br>Wang et al. (2017)<br>Wang et al. (2017) | Q<br>Q<br>Q | 80, 238<br>80, 239<br>80, 240 |
| MCM:MO2EOLA2OH<br>$C_3H_8O_3$<br>CSCSROFYRUZJJH-UHFFFAOYSA-N | $9.6\times10^3$<br>$9.6\times10^3$<br>$4.5\times10^2$ | | Wang et al. (2017)<br>Wang et al. (2017)<br>Wang et al. (2017) | Q<br>Q<br>Q | 80, 238<br>80, 239<br>80, 240 |



Table A3.9: Ethers (ROR) (...continued)

| Substance<br>Formula<br>(Trivial Name)<br>[CAS Registry Number]<br>InChIKey | $H_s^{cp}$<br>(at $T^\ominus$)<br>$\left[\dfrac{\mathrm{mol}}{\mathrm{m^3\,Pa}}\right]$ | $\dfrac{\mathrm{d}\ln H_s^{cp}}{\mathrm{d}(1/T)}$<br><br>[K] | Reference | Type | Note |
|---|---|---|---|---|---|
| MCM:MO2EOLAOOH<br>$C_3H_8O_4$<br>PFSIOEPXGRVTPO-UHFFFAOYSA-N | $8.1\times10^5$<br>$2.8\times10^4$<br>$6.9\times10^3$ | | Wang et al. (2017)<br>Wang et al. (2017)<br>Wang et al. (2017) | Q<br>Q<br>Q | 80, 238<br>80, 239<br>80, 240 |
| MCM:MO2EOLB2OH<br>$C_3H_8O_3$<br>ONSWVOSXVUHESJ-UHFFFAOYSA-N | $2.3\times10^4$<br>$2.0\times10^4$<br>$4.8\times10^4$ | | Wang et al. (2017)<br>Wang et al. (2017)<br>Wang et al. (2017) | Q<br>Q<br>Q | 80, 238<br>80, 239<br>80, 240 |
| MCM:MO2EOLBOOH<br>$C_3H_8O_4$<br>VULCDXDIAONDTM-UHFFFAOYSA-N | $6.0\times10^5$<br>$1.4\times10^5$<br>$2.5\times10^4$ | | Wang et al. (2017)<br>Wang et al. (2017)<br>Wang et al. (2017) | Q<br>Q<br>Q | 80, 238<br>80, 239<br>80, 240 |
| MCM:DIETETOH<br>$C_4H_{10}O_2$<br>CAFAOQIVXSSFSY-UHFFFAOYSA-N | $6.0$<br>$1.7\times10^1$<br>$1.1\times10^1$ | | Wang et al. (2017)<br>Wang et al. (2017)<br>Wang et al. (2017) | Q<br>Q<br>Q | 80, 238<br>80, 239<br>80, 240 |
| MCM:EOX2ETA2OH<br>$C_4H_{10}O_3$<br>WWVBRUMYFUDEJQ-UHFFFAOYSA-N | $8.0\times10^3$<br>$6.6\times10^3$<br>$4.6\times10^2$ | | Wang et al. (2017)<br>Wang et al. (2017)<br>Wang et al. (2017) | Q<br>Q<br>Q | 80, 238<br>80, 239<br>80, 240 |
| MCM:EOX2ETB2OH<br>$C_4H_{10}O_3$<br>RZYMXMZJVMXDRP-UHFFFAOYSA-N | $2.1\times10^4$<br>$8.9\times10^3$<br>$2.1\times10^4$ | | Wang et al. (2017)<br>Wang et al. (2017)<br>Wang et al. (2017) | Q<br>Q<br>Q | 80, 238<br>80, 239<br>80, 240 |
| MCM:EOX2OLAOOH<br>$C_4H_{10}O_4$<br>OSAXOMWTXDBXFG-UHFFFAOYSA-N | $6.3\times10^5$<br>$1.7\times10^4$<br>$4.1\times10^3$ | | Wang et al. (2017)<br>Wang et al. (2017)<br>Wang et al. (2017) | Q<br>Q<br>Q | 80, 238<br>80, 239<br>80, 240 |
| MCM:EOX2OLBOOH<br>$C_4H_{10}O_4$<br>IWDPPKCJRWMAAT-UHFFFAOYSA-N | $6.3\times10^5$<br>$6.2\times10^4$<br>$1.5\times10^4$ | | Wang et al. (2017)<br>Wang et al. (2017)<br>Wang et al. (2017) | Q<br>Q<br>Q | 80, 238<br>80, 239<br>80, 240 |
| MCM:H2C3OCOH<br>$C_4H_{10}O_3$<br>OAPWFUZUONIVKV-UHFFFAOYSA-N | $2.1\times10^4$<br>$1.0\times10^4$<br>$2.2\times10^4$ | | Wang et al. (2017)<br>Wang et al. (2017)<br>Wang et al. (2017) | Q<br>Q<br>Q | 80, 238<br>80, 239<br>80, 240 |
| MCM:H2C3OCOOH<br>$C_4H_{10}O_4$<br>ZZOFHQOVQLUGOX-UHFFFAOYSA-N | $6.3\times10^5$<br>$1.3\times10^5$<br>$1.6\times10^4$ | | Wang et al. (2017)<br>Wang et al. (2017)<br>Wang et al. (2017) | Q<br>Q<br>Q | 80, 238<br>80, 239<br>80, 240 |
| MCM:IPRMEETOH<br>$C_4H_{10}O_2$<br>BFSUQRCCKXZXEX-UHFFFAOYSA-N | $4.2$<br>$9.3$<br>$7.4$ | | Wang et al. (2017)<br>Wang et al. (2017)<br>Wang et al. (2017) | Q<br>Q<br>Q | 80, 238<br>80, 239<br>80, 240 |
| MCM:PR2OHMOOOH<br>$C_4H_{10}O_4$<br>BIISZUJLPYTCKS-UHFFFAOYSA-N | $7.6\times10^5$<br>$1.1\times10^4$<br>$2.7\times10^3$ | | Wang et al. (2017)<br>Wang et al. (2017)<br>Wang et al. (2017) | Q<br>Q<br>Q | 80, 238<br>80, 239<br>80, 240 |
| MCM:PROH2MOX<br>$C_4H_{10}O_3$<br>OEYNWAWWSZUGDU-UHFFFAOYSA-N | $8.9\times10^3$<br>$7.6\times10^3$<br>$9.3\times10^1$ | | Wang et al. (2017)<br>Wang et al. (2017)<br>Wang et al. (2017) | Q<br>Q<br>Q | 80, 238<br>80, 239<br>80, 240 |



Table A3.9: Ethers (ROR) (...continued)

| Substance Formula (Trivial Name) [CAS Registry Number] InChIKey | $H_s^{cp}$ (at $T^\ominus$) $\left[\dfrac{\mathrm{mol}}{\mathrm{m}^3\,\mathrm{Pa}}\right]$ | $\dfrac{\mathrm{d}\ln H_s^{cp}}{\mathrm{d}(1/T)}$ [K] | Reference | Type | Note |
|---|---|---|---|---|---|
| MCM:BOXMOH | 4.7 | | Wang et al. (2017) | Q | 80, 238 |
| $C_5H_{12}O_2$ | $1.5\times10^1$ | | Wang et al. (2017) | Q | 80, 239 |
| CRHLZRRTZDFDAJ-UHFFFAOYSA-N | $1.3\times10^1$ | | Wang et al. (2017) | Q | 80, 240 |
| MCM:EIPEOH | 3.4 | | Wang et al. (2017) | Q | 80, 238 |
| $C_5H_{12}O_2$ | 6.3 | | Wang et al. (2017) | Q | 80, 239 |
| MGMXTRZFWPWZFH-UHFFFAOYSA-N | 7.4 | | Wang et al. (2017) | Q | 80, 240 |
| MCM:HIEPOXB | $1.5\times10^8$ | | Wang et al. (2017) | Q | 80, 238 |
| $C_5H_{10}O_4$ | $1.8\times10^8$ | | Wang et al. (2017) | Q | 80, 239 |
| YTIKZIBPNMOYQX-UHFFFAOYSA-N | $5.6\times10^6$ | | Wang et al. (2017) | Q | 80, 240 |
| MCM:IEACO3H | $7.8\times10^5$ | | Wang et al. (2017) | Q | 80, 238 |
| $C_5H_8O_5$ | $3.6\times10^3$ | | Wang et al. (2017) | Q | 80, 239 |
| WBOLSVSGSKMCPN-UHFFFAOYSA-N | $1.7\times10^2$ | | Wang et al. (2017) | Q | 80, 240 |
| MCM:IECCO3H | $7.8\times10^5$ | | Wang et al. (2017) | Q | 80, 238 |
| $C_5H_8O_5$ | $2.7\times10^3$ | | Wang et al. (2017) | Q | 80, 239 |
| CEWHGLGNUIHIJU-UHFFFAOYSA-N | $1.1\times10^3$ | | Wang et al. (2017) | Q | 80, 240 |
| MCM:IEPOXA | $1.6\times10^4$ | | Wang et al. (2017) | Q | 80, 238 |
| $C_5H_{10}O_3$ | $9.3\times10^4$ | | Wang et al. (2017) | Q | 80, 239 |
| CIDUHKBATDRWPE-UHFFFAOYSA-N | $4.5\times10^2$ | | Wang et al. (2017) | Q | 80, 240 |
| MCM:IEPOXC | $1.6\times10^4$ | | Wang et al. (2017) | Q | 80, 238 |
| $C_5H_{10}O_3$ | $4.7\times10^4$ | | Wang et al. (2017) | Q | 80, 239 |
| REKLCZSNEUFIBP-UHFFFAOYSA-N | $1.1\times10^3$ | | Wang et al. (2017) | Q | 80, 240 |
| MCM:IPROC21OH | 5.6 | | Wang et al. (2017) | Q | 80, 238 |
| $C_5H_{12}O_2$ | $1.0\times10^1$ | | Wang et al. (2017) | Q | 80, 239 |
| PWLPTLCMMFJZIU-UHFFFAOYSA-N | 9.8 | | Wang et al. (2017) | Q | 80, 240 |
| MCM:MTBEAOH | 3.4 | | Wang et al. (2017) | Q | 80, 238 |
| $C_5H_{12}O_2$ | 7.4 | | Wang et al. (2017) | Q | 80, 239 |
| NHNNIMVRQJIZFW-UHFFFAOYSA-N | $2.7\times10^1$ | | Wang et al. (2017) | Q | 80, 240 |
| MCM:MTBEBOH | 4.8 | | Wang et al. (2017) | Q | 80, 238 |
| $C_5H_{12}O_2$ | $1.5\times10^1$ | | Wang et al. (2017) | Q | 80, 239 |
| VMPUAIZSESMILD-UHFFFAOYSA-N | 3.6 | | Wang et al. (2017) | Q | 80, 240 |
| MCM:BOX2E2OH | $5.5\times10^3$ | | Wang et al. (2017) | Q | 80, 238 |
| $C_6H_{14}O_3$ | $3.6\times10^3$ | | Wang et al. (2017) | Q | 80, 239 |
| WQHNZXURJISVCT-UHFFFAOYSA-N | $1.7\times10^2$ | | Wang et al. (2017) | Q | 80, 240 |
| MCM:BOX2OHAOOH | $4.1\times10^5$ | | Wang et al. (2017) | Q | 80, 238 |
| $C_6H_{14}O_4$ | $3.6\times10^4$ | | Wang et al. (2017) | Q | 80, 239 |
| XRZMTAXNHOGTME-UHFFFAOYSA-N | $1.0\times10^4$ | | Wang et al. (2017) | Q | 80, 240 |
| MCM:BOX2OHBOOH | $4.6\times10^5$ | | Wang et al. (2017) | Q | 80, 238 |
| $C_6H_{14}O_4$ | $8.1\times10^3$ | | Wang et al. (2017) | Q | 80, 239 |
| LCMMTUJPIXBSDS-UHFFFAOYSA-N | $2.2\times10^3$ | | Wang et al. (2017) | Q | 80, 240 |



**Rolf Sander: Compilation of Henry's law constants** 675

Table A3.9: Ethers (ROR) (...continued)

| Substance Formula (Trivial Name) [CAS Registry Number] InChIKey | $H_s^{cp}$ (at $T^\ominus$) $\left[\dfrac{\mathrm{mol}}{\mathrm{m}^3\,\mathrm{Pa}}\right]$ | $\dfrac{\mathrm{d}\ln H_s^{cp}}{\mathrm{d}(1/T)}$ [K] | Reference | Type | Note |
|---|---|---|---|---|---|
| MCM:BOXOHETOH | $1.5\times10^4$ | | Wang et al. (2017) | Q | 80, 238 |
| $C_6H_{14}O_3$ | $6.0\times10^3$ | | Wang et al. (2017) | Q | 80, 239 |
| QLKNFVRAYYHHDF-UHFFFAOYSA-N | $7.3\times10^3$ | | Wang et al. (2017) | Q | 80, 240 |
| MCM:DIIPRETOH | $3.2$ | | Wang et al. (2017) | Q | 80, 238 |
| $C_6H_{14}O_2$ | $3.5$ | | Wang et al. (2017) | Q | 80, 239 |
| FRKCZFUUIQCQPC-UHFFFAOYSA-N | $4.8$ | | Wang et al. (2017) | Q | 80, 240 |
| MCM:ETBEAOH | $3.7$ | | Wang et al. (2017) | Q | 80, 238 |
| $C_6H_{14}O_2$ | $1.0\times10^1$ | | Wang et al. (2017) | Q | 80, 239 |
| WOSZILCYMCIWFB-UHFFFAOYSA-N | $3.0$ | | Wang et al. (2017) | Q | 80, 240 |
| MCM:ETBEBOH | $3.2$ | | Wang et al. (2017) | Q | 80, 238 |
| $C_6H_{14}O_2$ | $4.1$ | | Wang et al. (2017) | Q | 80, 239 |
| DUNYNUFVLYAWTI-UHFFFAOYSA-N | $1.0\times10^1$ | | Wang et al. (2017) | Q | 80, 240 |
| MCM:ETBECOH | $3.7$ | | Wang et al. (2017) | Q | 80, 238 |
| $C_6H_{14}O_2$ | $8.5$ | | Wang et al. (2017) | Q | 80, 239 |
| BDLXTDLGTWNUFM-UHFFFAOYSA-N | $2.2\times10^1$ | | Wang et al. (2017) | Q | 80, 240 |
| MCM:IPROMC2OH | $6.5$ | | Wang et al. (2017) | Q | 80, 238 |
| $C_6H_{14}O_2$ | $1.3\times10^1$ | | Wang et al. (2017) | Q | 80, 239 |
| ZFEKANLLFQEKED-UHFFFAOYSA-N | $4.2$ | | Wang et al. (2017) | Q | 80, 240 |
| 1-butoxy-2-propanol | $5.0$ | | Wang et al. (2017) | Q | 80, 238 |
| $C_7H_{16}O_2$ | $1.7\times10^1$ | | Wang et al. (2017) | Q | 80, 239 |
| (MCM:BOX2PROL) | $8.7\times10^{-1}$ | | Wang et al. (2017) | Q | 80, 240 |
| [5131-66-8] | | | | | |
| RWNUSVWFHDHRCJ-UHFFFAOYSA-N | | | | | |
| MCM:BOXOHPROL | $1.4\times10^4$ | | Wang et al. (2017) | Q | 80, 238 |
| $C_7H_{16}O_3$ | $7.8\times10^3$ | | Wang et al. (2017) | Q | 80, 239 |
| ITIQJVOJGXCBQB-UHFFFAOYSA-N | $6.6\times10^3$ | | Wang et al. (2017) | Q | 80, 240 |
| MCM:BOXPOLAOOH | $3.8\times10^5$ | | Wang et al. (2017) | Q | 80, 238 |
| $C_7H_{16}O_4$ | $2.6\times10^4$ | | Wang et al. (2017) | Q | 80, 239 |
| JGCUFCZSCOGTSN-UHFFFAOYSA-N | $1.0\times10^3$ | | Wang et al. (2017) | Q | 80, 240 |
| MCM:BOXPOLBOOH | $4.3\times10^5$ | | Wang et al. (2017) | Q | 80, 238 |
| $C_7H_{16}O_4$ | $6.0\times10^3$ | | Wang et al. (2017) | Q | 80, 239 |
| AJCGQJRNRRPZGZ-UHFFFAOYSA-N | $9.1\times10^2$ | | Wang et al. (2017) | Q | 80, 240 |
| MCM:BOXPR2OH | $5.1\times10^3$ | | Wang et al. (2017) | Q | 80, 238 |
| $C_7H_{16}O_3$ | $2.9\times10^3$ | | Wang et al. (2017) | Q | 80, 239 |
| XKPKIGSYCLTAJO-UHFFFAOYSA-N | $4.4\times10^1$ | | Wang et al. (2017) | Q | 80, 240 |
| MCM:BCSOZOH | $1.1\times10^7$ | | Wang et al. (2017) | Q | 80, 238 |
| $C_{15}H_{26}O_5$ | $1.1\times10^5$ | | Wang et al. (2017) | Q | 80, 239 |
| JQRHKAPCWHNLMS-UHFFFAOYSA-N | $3.4\times10^6$ | | Wang et al. (2017) | Q | 80, 240 |
| MCM:BCSOZOOH | $8.5\times10^8$ | | Wang et al. (2017) | Q | 80, 238 |
| $C_{15}H_{26}O_6$ | $1.4\times10^5$ | | Wang et al. (2017) | Q | 80, 239 |
| VCKYKGHDZKOMMJ-UHFFFAOYSA-N | $5.0\times10^7$ | | Wang et al. (2017) | Q | 80, 240 |



Table A3.9: Ethers (ROR) (...continued)

| Substance Formula (Trivial Name) [CAS Registry Number] InChIKey | $H_s^{cp}$ (at $T^{\ominus}$) $\left[\dfrac{\text{mol}}{\text{m}^3\,\text{Pa}}\right]$ | $\dfrac{\text{d}\ln H_s^{cp}}{\text{d}(1/T)}$ [K] | Reference | Type | Note |
|---|---|---|---|---|---|
| MCM:MEMOXYCHO | 2.1 | | Wang et al. (2017) | Q | 80, 238 |
| $C_3H_6O_2$ | $2.0\times10^1$ | | Wang et al. (2017) | Q | 80, 239 |
| YSEFYOVWKJXNCH-UHFFFAOYSA-N | 3.0 | | Wang et al. (2017) | Q | 80, 240 |
| MCM:EOX2ETCHO | 2.0 | | Wang et al. (2017) | Q | 80, 238 |
| $C_4H_8O_2$ | 6.9 | | Wang et al. (2017) | Q | 80, 239 |
| IAHZBRPNDIVNNR-UHFFFAOYSA-N | 2.3 | | Wang et al. (2017) | Q | 80, 240 |
| MCM:EPXC4DIAL | $6.5\times10^3$ | | Wang et al. (2017) | Q | 80, 238 |
| $C_4H_4O_3$ | $7.1\times10^3$ | | Wang et al. (2017) | Q | 80, 239 |
| IRJHVNZVWOCVLV-UHFFFAOYSA-N | 7.8 | | Wang et al. (2017) | Q | 80, 240 |
| MCM:EPXDLCO3H | $7.1\times10^6$ | | Wang et al. (2017) | Q | 80, 238 |
| $C_4H_4O_5$ | $9.6\times10^4$ | | Wang et al. (2017) | Q | 80, 239 |
| OKYIQGCZHSMTGV-UHFFFAOYSA-N | $7.6\times10^1$ | | Wang et al. (2017) | Q | 80, 240 |
| MCM:EPXMC4DIAL | $3.5\times10^3$ | | Wang et al. (2017) | Q | 80, 238 |
| $C_5H_6O_3$ | $1.2\times10^3$ | | Wang et al. (2017) | Q | 80, 239 |
| SYCYSIWUSJFZTN-UHFFFAOYSA-N | 7.4 | | Wang et al. (2017) | Q | 80, 240 |
| MCM:EPXMDLCO3H | $3.8\times10^6$ | | Wang et al. (2017) | Q | 80, 238 |
| $C_5H_6O_5$ | $3.1\times10^4$ | | Wang et al. (2017) | Q | 80, 239 |
| LLYFBEXLHOFXFI-UHFFFAOYSA-N | $2.8\times10^2$ | | Wang et al. (2017) | Q | 80, 240 |
| MCM:MTBEBCHO | 1.1 | | Wang et al. (2017) | Q | 80, 238 |
| $C_5H_{10}O_2$ | 1.5 | | Wang et al. (2017) | Q | 80, 239 |
| HSCUZOQCNBPBST-UHFFFAOYSA-N | $5.8\times10^{-1}$ | | Wang et al. (2017) | Q | 80, 240 |
| MCM:BOX2ECHO | 1.2 | | Wang et al. (2017) | Q | 80, 238 |
| $C_6H_{12}O_2$ | 6.5 | | Wang et al. (2017) | Q | 80, 239 |
| RPLPGIHCAYAYKX-UHFFFAOYSA-N | 1.1 | | Wang et al. (2017) | Q | 80, 240 |
| MCM:BZEMUCCO3H | $2.1\times10^7$ | | Wang et al. (2017) | Q | 80, 238 |
| $C_6H_6O_5$ | $1.1\times10^5$ | | Wang et al. (2017) | Q | 80, 239 |
| LTCNUSVDDXPSIF-UHFFFAOYSA-N | $1.2\times10^3$ | | Wang et al. (2017) | Q | 80, 240 |
| MCM:BZEPOXMUC | $1.9\times10^4$ | | Wang et al. (2017) | Q | 80, 238 |
| $C_6H_6O_3$ | $5.4\times10^3$ | | Wang et al. (2017) | Q | 80, 239 |
| NQHJMOLWTXQPLS-UHFFFAOYSA-N | $1.8\times10^3$ | | Wang et al. (2017) | Q | 80, 240 |
| MCM:EPXM2C4DAL | $2.0\times10^3$ | | Wang et al. (2017) | Q | 80, 238 |
| $C_6H_8O_3$ | $3.0\times10^2$ | | Wang et al. (2017) | Q | 80, 239 |
| QTKFOEFUZGORGI-UHFFFAOYSA-N | $4.5\times10^1$ | | Wang et al. (2017) | Q | 80, 240 |
| MCM:EPXM2DCO3H | $2.2\times10^6$ | | Wang et al. (2017) | Q | 80, 238 |
| $C_6H_8O_5$ | $6.8\times10^3$ | | Wang et al. (2017) | Q | 80, 239 |
| CFBRPHMFNHCZFV-UHFFFAOYSA-N | $1.6\times10^1$ | | Wang et al. (2017) | Q | 80, 240 |
| MCM:ETBEACHO | $9.8\times10^{-1}$ | | Wang et al. (2017) | Q | 80, 238 |
| $C_6H_{12}O_2$ | 1.2 | | Wang et al. (2017) | Q | 80, 239 |
| HHIYJPQUMUPZIZ-UHFFFAOYSA-N | $4.5\times10^{-1}$ | | Wang et al. (2017) | Q | 80, 240 |





Table A3.9: Ethers (ROR) (... continued)

| Substance Formula (Trivial Name) [CAS Registry Number] InChIKey | $H_s^{cp}$ (at $T^\ominus$) $\left[\dfrac{\mathrm{mol}}{\mathrm{m^3\,Pa}}\right]$ | $\dfrac{\mathrm{d}\ln H_s^{cp}}{\mathrm{d}(1/T)}$ [K] | Reference | Type | Note |
|---|---|---|---|---|---|
| MCM:ETBECCHO | $9.8\times10^{-1}$ | | Wang et al. (2017) | Q | 80, 238 |
| $C_6H_{12}O_2$ | 3.5 | | Wang et al. (2017) | Q | 80, 239 |
| SMUYZOMGNHBYHU-UHFFFAOYSA-N | 6.8 | | Wang et al. (2017) | Q | 80, 240 |
| MCM:IPROMCCHO | 1.6 | | Wang et al. (2017) | Q | 80, 238 |
| $C_6H_{12}O_2$ | 3.3 | | Wang et al. (2017) | Q | 80, 239 |
| AAPDVSFWBQNMLJ-UHFFFAOYSA-N | 1.2 | | Wang et al. (2017) | Q | 80, 240 |
| MCM:EPXMEC4DAL | $1.6\times10^3$ | | Wang et al. (2017) | Q | 80, 238 |
| $C_7H_{10}O_3$ | $2.8\times10^2$ | | Wang et al. (2017) | Q | 80, 239 |
| MOJUYSNCEBTQMW-UHFFFAOYSA-N | 2.5 | | Wang et al. (2017) | Q | 80, 240 |
| MCM:EPXMEDCO3H | $1.7\times10^6$ | | Wang et al. (2017) | Q | 80, 238 |
| $C_7H_{10}O_5$ | $3.7\times10^3$ | | Wang et al. (2017) | Q | 80, 239 |
| IBELTJZUJIPVRI-UHFFFAOYSA-N | 5.6 | | Wang et al. (2017) | Q | 80, 240 |
| MCM:OXYEPOXMUC | $5.0\times10^3$ | | Wang et al. (2017) | Q | 80, 238 |
| $C_8H_{10}O_3$ | $6.8\times10^2$ | | Wang et al. (2017) | Q | 80, 239 |
| QLTFECLMPRWGQQ-UHFFFAOYSA-N | $1.1\times10^3$ | | Wang et al. (2017) | Q | 80, 240 |
| MCM:OXYMUCCO3H | $6.2\times10^6$ | | Wang et al. (2017) | Q | 80, 238 |
| $C_8H_{10}O_5$ | $6.6\times10^3$ | | Wang et al. (2017) | Q | 80, 239 |
| PBSKNYKKTPMDFL-UHFFFAOYSA-N | $8.5\times10^2$ | | Wang et al. (2017) | Q | 80, 240 |
| MCM:OETLMUCO3H | $5.0\times10^6$ | | Wang et al. (2017) | Q | 80, 238 |
| $C_9H_{12}O_5$ | $4.0\times10^3$ | | Wang et al. (2017) | Q | 80, 239 |
| UVJSWMTUJCHEOZ-UHFFFAOYSA-N | $5.0\times10^2$ | | Wang et al. (2017) | Q | 80, 240 |
| MCM:OETLPOXMUC | $4.6\times10^3$ | | Wang et al. (2017) | Q | 80, 238 |
| $C_9H_{12}O_3$ | $4.6\times10^2$ | | Wang et al. (2017) | Q | 80, 239 |
| NORSPZZKBNMHEY-UHFFFAOYSA-N | $5.9\times10^2$ | | Wang et al. (2017) | Q | 80, 240 |
| MCM:IEACHO | $7.1\times10^2$ | | Wang et al. (2017) | Q | 80, 238 |
| $C_5H_8O_3$ | $4.2\times10^3$ | | Wang et al. (2017) | Q | 80, 239 |
| SXTKPSXTIDLSQV-UHFFFAOYSA-N | $3.9\times10^1$ | | Wang et al. (2017) | Q | 80, 240 |
| MCM:IECCHO | $7.1\times10^2$ | | Wang et al. (2017) | Q | 80, 238 |
| $C_5H_8O_3$ | $1.9\times10^3$ | | Wang et al. (2017) | Q | 80, 239 |
| FUNIHXJAXNKVQS-UHFFFAOYSA-N | $2.3\times10^1$ | | Wang et al. (2017) | Q | 80, 240 |
| MCM:BZEMUCOH | $1.0\times10^9$ | | Wang et al. (2017) | Q | 80, 238 |
| $C_6H_8O_5$ | $1.7\times10^9$ | | Wang et al. (2017) | Q | 80, 239 |
| HYRDKGAOZPVDDL-UHFFFAOYSA-N | $1.1\times10^5$ | | Wang et al. (2017) | Q | 80, 240 |
| MCM:BZEMUCOOH | $7.6\times10^{10}$ | | Wang et al. (2017) | Q | 80, 238 |
| $C_6H_8O_6$ | $1.2\times10^9$ | | Wang et al. (2017) | Q | 80, 239 |
| YPGXWAYLBNMEJL-UHFFFAOYSA-N | $1.4\times10^6$ | | Wang et al. (2017) | Q | 80, 240 |
| MCM:OXYMUCOH | $3.1\times10^8$ | | Wang et al. (2017) | Q | 80, 238 |
| $C_8H_{12}O_5$ | $1.3\times10^8$ | | Wang et al. (2017) | Q | 80, 239 |
| SKTLSFGOHJFJLD-UHFFFAOYSA-N | $9.3\times10^3$ | | Wang et al. (2017) | Q | 80, 240 |



Table A3.9: Ethers (ROR) (...continued)

| Substance Formula (Trivial Name) [CAS Registry Number] InChIKey | $H_s^{cp}$ (at $T^{\ominus}$) $\left[\dfrac{\mathrm{mol}}{\mathrm{m^3\,Pa}}\right]$ | $\dfrac{\mathrm{d}\ln H_s^{cp}}{\mathrm{d}(1/T)}$ [K] | Reference | Type | Note |
|---|---|---|---|---|---|
| MCM:OXYMUCOOH $C_8H_{12}O_6$ ZBCSCRLAPOVYAB-UHFFFAOYSA-N | $4.6\times10^{11}$ $2.9\times10^{8}$ $1.7\times10^{4}$ | | Wang et al. (2017) Wang et al. (2017) Wang et al. (2017) | Q Q Q | 80, 238 80, 239 80, 240 |
| MCM:OETLMUCOH $C_9H_{14}O_5$ BKCWOZADWPGCRT-UHFFFAOYSA-N | $2.6\times10^{8}$ $7.4\times10^{7}$ $4.5\times10^{3}$ | | Wang et al. (2017) Wang et al. (2017) Wang et al. (2017) | Q Q Q | 80, 238 80, 239 80, 240 |
| MCM:OETLMUCOOH $C_9H_{14}O_6$ OMPONQJDZISYFQ-UHFFFAOYSA-N | $4.2\times10^{11}$ $1.7\times10^{8}$ $1.1\times10^{3}$ | | Wang et al. (2017) Wang et al. (2017) Wang et al. (2017) | Q Q Q | 80, 238 80, 239 80, 240 |
| MCM:PRONEMOOOH $C_4H_8O_4$ WKDIYCKWLNCFHJ-UHFFFAOYSA-N | $1.2\times10^{5}$ $8.1\times10^{2}$ $2.0\times10^{2}$ | | Wang et al. (2017) Wang et al. (2017) Wang et al. (2017) | Q Q Q | 80, 238 80, 239 80, 240 |
| MCM:PRONEMOX $C_4H_8O_2$ CUZLJOLBIRPEFB-UHFFFAOYSA-N | $1.4$ $1.4\times10^{1}$ $8.1$ | | Wang et al. (2017) Wang et al. (2017) Wang et al. (2017) | Q Q Q | 80, 238 80, 239 80, 240 |
| MCM:EPXKTMCO3H $C_6H_8O_5$ ABDSVLWFPNKKBV-UHFFFAOYSA-N | $2.6\times10^{6}$ $3.2\times10^{4}$ $4.5\times10^{2}$ | | Wang et al. (2017) Wang et al. (2017) Wang et al. (2017) | Q Q Q | 80, 238 80, 239 80, 240 |
| MCM:EPXMKTCO3H $C_6H_8O_5$ CAGQARJHZCESJI-UHFFFAOYSA-N | $2.6\times10^{6}$ $3.2\times10^{4}$ $1.3\times10^{3}$ | | Wang et al. (2017) Wang et al. (2017) Wang et al. (2017) | Q Q Q | 80, 238 80, 239 80, 240 |
| MCM:BOXPROBOOH $C_7H_{14}O_4$ YHFJWWLEGMYFFZ-UHFFFAOYSA-N | $7.1\times10^{4}$ $2.0\times10^{2}$ $3.6\times10^{1}$ | | Wang et al. (2017) Wang et al. (2017) Wang et al. (2017) | Q Q Q | 80, 238 80, 239 80, 240 |
| MCM:BOXPRONE $C_7H_{14}O_2$ UYNCDYOFUJEUQN-UHFFFAOYSA-N | $8.1\times10^{-1}$ $8.0$ $2.0$ | | Wang et al. (2017) Wang et al. (2017) Wang et al. (2017) | Q Q Q | 80, 238 80, 239 80, 240 |
| MCM:BOXPROOOH $C_7H_{14}O_4$ YUTHBWBUPOPMPG-UHFFFAOYSA-N | $7.1\times10^{4}$ $6.3\times10^{3}$ $2.9\times10^{2}$ | | Wang et al. (2017) Wang et al. (2017) Wang et al. (2017) | Q Q Q | 80, 238 80, 239 80, 240 |
| MCM:EPXEKTCO3H $C_7H_{10}O_5$ PGCJNBUAAIEYIA-UHFFFAOYSA-N | $2.0\times10^{6}$ $2.4\times10^{4}$ $2.4\times10^{2}$ | | Wang et al. (2017) Wang et al. (2017) Wang et al. (2017) | Q Q Q | 80, 238 80, 239 80, 240 |
| MCM:TLEMUCCO3H $C_7H_8O_5$ RISGILSSNRWCFR-UHFFFAOYSA-N | $1.4\times10^{7}$ $2.0\times10^{5}$ $5.9\times10^{3}$ | | Wang et al. (2017) Wang et al. (2017) Wang et al. (2017) | Q Q Q | 80, 238 80, 239 80, 240 |
| MCM:EBZMUCCO3H $C_8H_{10}O_5$ GTEVSPXWKYUSQE-UHFFFAOYSA-N | $1.1\times10^{7}$ $9.6\times10^{4}$ $1.0\times10^{4}$ | | Wang et al. (2017) Wang et al. (2017) Wang et al. (2017) | Q Q Q | 80, 238 80, 239 80, 240 |





Table A3.9: Ethers (ROR) (...continued)

| Substance Formula (Trivial Name) [CAS Registry Number] InChIKey | $H_s^{cp}$ (at $T^\ominus$) $\left[\dfrac{\text{mol}}{\text{m}^3\,\text{Pa}}\right]$ | $\dfrac{\text{d}\ln H_s^{cp}}{\text{d}(1/T)}$ [K] | Reference | Type | Note |
|---|---|---|---|---|---|
| MCM:MXYMUCCO3H | $7.6\times10^6$ | | Wang et al. (2017) | Q | 80, 238 |
| $C_8H_{10}O_5$ | $3.6\times10^4$ | | Wang et al. (2017) | Q | 80, 239 |
| ROKAXOITEMHIEK-UHFFFAOYSA-N | $3.8\times10^4$ | | Wang et al. (2017) | Q | 80, 240 |
| MCM:PXYMUCCO3H | $7.6\times10^6$ | | Wang et al. (2017) | Q | 80, 238 |
| $C_8H_{10}O_5$ | $5.5\times10^4$ | | Wang et al. (2017) | Q | 80, 239 |
| AYUPQGRQZSTAMO-UHFFFAOYSA-N | $1.9\times10^4$ | | Wang et al. (2017) | Q | 80, 240 |
| MCM:IPBZMUCO3H | $1.0\times10^7$ | | Wang et al. (2017) | Q | 80, 238 |
| $C_9H_{12}O_5$ | $5.6\times10^4$ | | Wang et al. (2017) | Q | 80, 239 |
| SKJBPGYNTSYQCQ-UHFFFAOYSA-N | $9.3\times10^3$ | | Wang et al. (2017) | Q | 80, 240 |
| MCM:METLMUCO3H | $6.0\times10^6$ | | Wang et al. (2017) | Q | 80, 238 |
| $C_9H_{12}O_5$ | $1.7\times10^4$ | | Wang et al. (2017) | Q | 80, 239 |
| CXRRYJPLLOUVIM-UHFFFAOYSA-N | $1.2\times10^3$ | | Wang et al. (2017) | Q | 80, 240 |
| MCM:PBZMUCCO3H | $9.8\times10^6$ | | Wang et al. (2017) | Q | 80, 238 |
| $C_9H_{12}O_5$ | $5.4\times10^4$ | | Wang et al. (2017) | Q | 80, 239 |
| ZGGPGQBINYKWGU-UHFFFAOYSA-N | $1.3\times10^4$ | | Wang et al. (2017) | Q | 80, 240 |
| MCM:PETLMUCO3H | $6.0\times10^6$ | | Wang et al. (2017) | Q | 80, 238 |
| $C_9H_{12}O_5$ | $2.6\times10^4$ | | Wang et al. (2017) | Q | 80, 239 |
| KWWJCAVGGMMMRV-UHFFFAOYSA-N | $7.8\times10^3$ | | Wang et al. (2017) | Q | 80, 240 |
| MCM:TM123OXMUC | $4.2\times10^3$ | | Wang et al. (2017) | Q | 80, 238 |
| $C_9H_{12}O_3$ | $3.9\times10^3$ | | Wang et al. (2017) | Q | 80, 239 |
| XYHNNFZVTWOCNN-UHFFFAOYSA-N | $1.7\times10^4$ | | Wang et al. (2017) | Q | 80, 240 |
| MCM:TM124MUO3H | $4.2\times10^6$ | | Wang et al. (2017) | Q | 80, 238 |
| $C_9H_{12}O_5$ | $1.2\times10^4$ | | Wang et al. (2017) | Q | 80, 239 |
| FTWAFKXUKWKGAJ-UHFFFAOYSA-N | $2.3\times10^3$ | | Wang et al. (2017) | Q | 80, 240 |
| MCM:TM135MUO3H | $5.3\times10^6$ | | Wang et al. (2017) | Q | 80, 238 |
| $C_9H_{12}O_5$ | $3.4\times10^4$ | | Wang et al. (2017) | Q | 80, 239 |
| JNFPQQOSCDIIIK-UHFFFAOYSA-N | $6.3\times10^2$ | | Wang et al. (2017) | Q | 80, 240 |
| MCM:DMEBMUO3H | $4.1\times10^6$ | | Wang et al. (2017) | Q | 80, 238 |
| $C_{10}H_{14}O_5$ | $1.7\times10^4$ | | Wang et al. (2017) | Q | 80, 239 |
| JWBQDZWURHDWCA-UHFFFAOYSA-N | $1.7\times10^2$ | | Wang et al. (2017) | Q | 80, 240 |
| MCM:DETLMUO3H | $3.6\times10^6$ | | Wang et al. (2017) | Q | 80, 238 |
| $C_{11}H_{16}O_5$ | $1.1\times10^4$ | | Wang et al. (2017) | Q | 80, 239 |
| QEEAHFHJZRPBGG-UHFFFAOYSA-N | $1.1\times10^2$ | | Wang et al. (2017) | Q | 80, 240 |
| MCM:BCKSOZ | $3.7\times10^3$ | | Wang et al. (2017) | Q | 80, 238 |
| $C_{14}H_{22}O_4$ | $1.4\times10^2$ | | Wang et al. (2017) | Q | 80, 239 |
| JTZKRSQXPBULRM-UHFFFAOYSA-N | $1.2\times10^4$ | | Wang et al. (2017) | Q | 80, 240 |
| MCM:PRONEMOXOH | $2.4\times10^2$ | | Wang et al. (2017) | Q | 80, 238 |
| $C_4H_8O_3$ | $1.4\times10^3$ | | Wang et al. (2017) | Q | 80, 239 |
| RETWRLMOZHRKHX-UHFFFAOYSA-N | 6.6 | | Wang et al. (2017) | Q | 80, 240 |



Table A3.9: Ethers (ROR) (...continued)

| Substance<br>Formula<br>(Trivial Name)<br>[CAS Registry Number]<br>InChIKey | $H_s^{cp}$<br>(at $T^{\ominus}$)<br>$\left[\dfrac{\text{mol}}{\text{m}^3\,\text{Pa}}\right]$ | $\dfrac{\text{d}\ln H_s^{cp}}{\text{d}(1/T)}$<br><br>[K] | Reference | Type | Note |
|---|---|---|---|---|---|
| MCM:BOXPRONBOH<br>$C_7H_{14}O_3$<br>ANPLAICHKIQBPV-UHFFFAOYSA-N | $1.2\times10^2$<br>$3.2\times10^2$<br>$1.4$ | | Wang et al. (2017)<br>Wang et al. (2017)<br>Wang et al. (2017) | Q<br>Q<br>Q | 80, 238<br>80, 239<br>80, 240 |
| MCM:BOXPRONOH<br>$C_7H_{14}O_3$<br>NQJNHGUTJISDHM-UHFFFAOYSA-N | $2.7\times10^3$<br>$1.0\times10^4$<br>$9.3\times10^2$ | | Wang et al. (2017)<br>Wang et al. (2017)<br>Wang et al. (2017) | Q<br>Q<br>Q | 80, 238<br>80, 239<br>80, 240 |
| MCM:TM123MUCCO<br>$C_9H_{12}O_5$<br>ZVWGUKJQFRZUIG-UHFFFAOYSA-N | $2.6\times10^7$<br>$4.1\times10^7$<br>$1.4\times10^3$ | | Wang et al. (2017)<br>Wang et al. (2017)<br>Wang et al. (2017) | Q<br>Q<br>Q | 80, 238<br>80, 239<br>80, 240 |
| MCM:TM123MUCOH<br>$C_9H_{14}O_5$<br>JUUFKUYJQYFRGA-UHFFFAOYSA-N | $2.6\times10^8$<br>$3.7\times10^8$<br>$1.1\times10^5$ | | Wang et al. (2017)<br>Wang et al. (2017)<br>Wang et al. (2017) | Q<br>Q<br>Q | 80, 238<br>80, 239<br>80, 240 |
| MCM:TM123MUOOH<br>$C_9H_{14}O_6$<br>LLVDHGQYZCXGKH-UHFFFAOYSA-N | $1.7\times10^{10}$<br>$2.7\times10^8$<br>$1.9\times10^5$ | | Wang et al. (2017)<br>Wang et al. (2017)<br>Wang et al. (2017) | Q<br>Q<br>Q | 80, 238<br>80, 239<br>80, 240 |
| MCM:EPXALMKT<br>$C_6H_8O_3$<br>QIJMAVIYYVSLPO-UHFFFAOYSA-N | $2.1\times10^3$<br>$9.8\times10^2$<br>$4.2\times10^1$ | | Wang et al. (2017)<br>Wang et al. (2017)<br>Wang et al. (2017) | Q<br>Q<br>Q | 80, 238<br>80, 239<br>80, 240 |
| MCM:EPXMALKT<br>$C_6H_8O_3$<br>MWBGBWUOLRJCDU-UHFFFAOYSA-N | $2.1\times10^3$<br>$9.8\times10^2$<br>$3.3\times10^2$ | | Wang et al. (2017)<br>Wang et al. (2017)<br>Wang et al. (2017) | Q<br>Q<br>Q | 80, 238<br>80, 239<br>80, 240 |
| MCM:EPXEALKT<br>$C_7H_{10}O_3$<br>HQSBJPSKMKUTIT-UHFFFAOYSA-N | $2.0\times10^3$<br>$1.0\times10^3$<br>$9.3\times10^1$ | | Wang et al. (2017)<br>Wang et al. (2017)<br>Wang et al. (2017) | Q<br>Q<br>Q | 80, 238<br>80, 239<br>80, 240 |
| MCM:EBZPOXMUC<br>$C_8H_{10}O_3$<br>UTCVUMOGNJSPDS-UHFFFAOYSA-N | $1.0\times10^4$<br>$9.1\times10^3$<br>$6.3\times10^2$ | | Wang et al. (2017)<br>Wang et al. (2017)<br>Wang et al. (2017) | Q<br>Q<br>Q | 80, 238<br>80, 239<br>80, 240 |
| MCM:MXYEPOXMUC<br>$C_8H_{10}O_3$<br>VLMKIGFKBHEGBU-UHFFFAOYSA-N | $6.2\times10^3$<br>$3.5\times10^3$<br>$1.2\times10^3$ | | Wang et al. (2017)<br>Wang et al. (2017)<br>Wang et al. (2017) | Q<br>Q<br>Q | 80, 238<br>80, 239<br>80, 240 |
| MCM:PXYEPOXMUC<br>$C_8H_{10}O_3$<br>LGKLIBFTBUCZPW-UHFFFAOYSA-N | $6.2\times10^3$<br>$5.4\times10^3$<br>$1.7\times10^3$ | | Wang et al. (2017)<br>Wang et al. (2017)<br>Wang et al. (2017) | Q<br>Q<br>Q | 80, 238<br>80, 239<br>80, 240 |
| MCM:IPBZPOXMUC<br>$C_9H_{12}O_3$<br>NUCYHGVQTCVYLO-UHFFFAOYSA-N | $9.6\times10^3$<br>$6.6\times10^3$<br>$4.1\times10^2$ | | Wang et al. (2017)<br>Wang et al. (2017)<br>Wang et al. (2017) | Q<br>Q<br>Q | 80, 238<br>80, 239<br>80, 240 |
| MCM:METLPOXMUC<br>$C_9H_{12}O_3$<br>SWHMTTAIKVELRE-UHFFFAOYSA-N | $5.5\times10^3$<br>$1.9\times10^3$<br>$4.8\times10^2$ | | Wang et al. (2017)<br>Wang et al. (2017)<br>Wang et al. (2017) | Q<br>Q<br>Q | 80, 238<br>80, 239<br>80, 240 |



Table A3.9: Ethers (ROR) (. . . continued)

| Substance<br>Formula<br>(Trivial Name)<br>[CAS Registry Number]<br>InChIKey | $H_s^{cp}$<br>(at $T^\ominus$)<br>$\left[\dfrac{\mathrm{mol}}{\mathrm{m^3\,Pa}}\right]$ | $\dfrac{\mathrm{d}\ln H_s^{cp}}{\mathrm{d}(1/T)}$<br><br>[K] | Reference | Type | Note |
|---|---|---|---|---|---|
| MCM:PBZPOXMUC | $8.0\times10^3$ | | Wang et al. (2017) | Q | 80, 238 |
| $C_9H_{12}O_3$ | $6.0\times10^3$ | | Wang et al. (2017) | Q | 80, 239 |
| AHKQGNMGHQLOPP-UHFFFAOYSA-N | $3.2\times10^2$ | | Wang et al. (2017) | Q | 80, 240 |
| MCM:PETLPOXMUC | $5.5\times10^3$ | | Wang et al. (2017) | Q | 80, 238 |
| $C_9H_{12}O_3$ | $3.0\times10^3$ | | Wang et al. (2017) | Q | 80, 239 |
| UTBNMXYREJYVNQ-UHFFFAOYSA-N | $6.0\times10^2$ | | Wang et al. (2017) | Q | 80, 240 |
| MCM:TM124OXMUC | $3.4\times10^3$ | | Wang et al. (2017) | Q | 80, 238 |
| $C_9H_{12}O_3$ | $1.3\times10^3$ | | Wang et al. (2017) | Q | 80, 239 |
| HFTMUUPKJAPOHL-UHFFFAOYSA-N | $2.0\times10^3$ | | Wang et al. (2017) | Q | 80, 240 |
| MCM:TM135OXMUC | $4.2\times10^3$ | | Wang et al. (2017) | Q | 80, 238 |
| $C_9H_{12}O_3$ | $3.7\times10^3$ | | Wang et al. (2017) | Q | 80, 239 |
| JFCWARPMSNNZLY-UHFFFAOYSA-N | $3.0\times10^2$ | | Wang et al. (2017) | Q | 80, 240 |
| MCM:DMEBPOXMUC | $3.7\times10^3$ | | Wang et al. (2017) | Q | 80, 238 |
| $C_{10}H_{14}O_3$ | $2.0\times10^3$ | | Wang et al. (2017) | Q | 80, 239 |
| VGOICOHBXPGRQT-UHFFFAOYSA-N | $1.3\times10^2$ | | Wang et al. (2017) | Q | 80, 240 |
| MCM:DETLPOXMUC | $3.0\times10^3$ | | Wang et al. (2017) | Q | 80, 238 |
| $C_{11}H_{16}O_3$ | $1.4\times10^3$ | | Wang et al. (2017) | Q | 80, 239 |
| VPGNNECRJBEQEI-UHFFFAOYSA-N | $6.5\times10^1$ | | Wang et al. (2017) | Q | 80, 240 |
| MCM:BZEMUCCO | $1.0\times10^8$ | | Wang et al. (2017) | Q | 80, 238 |
| $C_6H_6O_5$ | $1.1\times10^8$ | | Wang et al. (2017) | Q | 80, 239 |
| KQXCODNTIAPVKL-UHFFFAOYSA-N | $3.0\times10^3$ | | Wang et al. (2017) | Q | 80, 240 |
| MCM:TLEMUCCO | $3.7\times10^8$ | | Wang et al. (2017) | Q | 80, 238 |
| $C_7H_8O_5$ | $1.7\times10^7$ | | Wang et al. (2017) | Q | 80, 239 |
| AJWOUZMMNQLKHW-UHFFFAOYSA-N | $3.0\times10^3$ | | Wang et al. (2017) | Q | 80, 240 |
| MCM:TLEMUCOH | $6.8\times10^8$ | | Wang et al. (2017) | Q | 80, 238 |
| $C_7H_{10}O_5$ | $1.5\times10^9$ | | Wang et al. (2017) | Q | 80, 239 |
| WPTVRQIFDQMAMS-UHFFFAOYSA-N | $5.1\times10^4$ | | Wang et al. (2017) | Q | 80, 240 |
| MCM:TLEMUCOOH | $1.0\times10^{12}$ | | Wang et al. (2017) | Q | 80, 238 |
| $C_7H_{10}O_6$ | $4.8\times10^9$ | | Wang et al. (2017) | Q | 80, 239 |
| DVPKQTIPCHKXBD-UHFFFAOYSA-N | $4.8\times10^4$ | | Wang et al. (2017) | Q | 80, 240 |
| MCM:EBZMUCCO | $3.5\times10^8$ | | Wang et al. (2017) | Q | 80, 238 |
| $C_8H_{10}O_5$ | $7.6\times10^6$ | | Wang et al. (2017) | Q | 80, 239 |
| MYHWFJZUBUHIGJ-UHFFFAOYSA-N | $2.2\times10^3$ | | Wang et al. (2017) | Q | 80, 240 |
| MCM:EBZMUCOH | $5.6\times10^8$ | | Wang et al. (2017) | Q | 80, 238 |
| $C_8H_{12}O_5$ | $1.0\times10^9$ | | Wang et al. (2017) | Q | 80, 239 |
| VKTUBYIQTACODT-UHFFFAOYSA-N | $3.0\times10^4$ | | Wang et al. (2017) | Q | 80, 240 |
| MCM:EBZMUCOOH | $9.1\times10^{11}$ | | Wang et al. (2017) | Q | 80, 238 |
| $C_8H_{12}O_6$ | $3.3\times10^9$ | | Wang et al. (2017) | Q | 80, 239 |
| KWCWGZCSJYJCAS-UHFFFAOYSA-N | $8.1\times10^3$ | | Wang et al. (2017) | Q | 80, 240 |



Table A3.9: Ethers (ROR) (. . . continued)

| Substance Formula (Trivial Name) [CAS Registry Number] InChIKey | $H_s^{cp}$ (at $T^{\ominus}$) $\left[\dfrac{\text{mol}}{\text{m}^3\,\text{Pa}}\right]$ | $\dfrac{\text{d}\ln H_s^{cp}}{\text{d}(1/T)}$ [K] | Reference | Type | Note |
|---|---|---|---|---|---|
| MCM:MXYMUCCO $C_8H_{10}O_5$ KJOIECUJZBFUMR-UHFFFAOYSA-N | $2.1\times10^8$ $3.3\times10^6$ $9.8\times10^2$ | | Wang et al. (2017) Wang et al. (2017) Wang et al. (2017) | Q Q Q | 80, 238 80, 239 80, 240 |
| MCM:MXYMUCOH $C_8H_{12}O_5$ UZLLCOIUEBQPHF-UHFFFAOYSA-N | $3.9\times10^8$ $3.9\times10^8$ $4.2\times10^4$ | | Wang et al. (2017) Wang et al. (2017) Wang et al. (2017) | Q Q Q | 80, 238 80, 239 80, 240 |
| MCM:MXYMUCOOH $C_8H_{12}O_6$ ZYWDOJNDZBPDDO-UHFFFAOYSA-N | $5.6\times10^{11}$ $1.2\times10^9$ $8.3\times10^3$ | | Wang et al. (2017) Wang et al. (2017) Wang et al. (2017) | Q Q Q | 80, 238 80, 239 80, 240 |
| MCM:OXYMUCCO $C_8H_{10}O_5$ RJHLIQPRSGXWHC-UHFFFAOYSA-N | $1.7\times10^8$ $1.9\times10^6$ $2.6\times10^2$ | | Wang et al. (2017) Wang et al. (2017) Wang et al. (2017) | Q Q Q | 80, 238 80, 239 80, 240 |
| MCM:PXYMUCCO $C_8H_{10}O_5$ UICQALDYFNTPHO-UHFFFAOYSA-N | $2.1\times10^8$ $4.9\times10^6$ $1.2\times10^3$ | | Wang et al. (2017) Wang et al. (2017) Wang et al. (2017) | Q Q Q | 80, 238 80, 239 80, 240 |
| MCM:PXYMUCOH $C_8H_{12}O_5$ MIBKDAKBVPRIGB-UHFFFAOYSA-N | $3.9\times10^8$ $5.1\times10^8$ $1.1\times10^4$ | | Wang et al. (2017) Wang et al. (2017) Wang et al. (2017) | Q Q Q | 80, 238 80, 239 80, 240 |
| MCM:PXYMUCOOH $C_8H_{12}O_6$ GSJPWDRXFYTPMH-UHFFFAOYSA-N | $5.6\times10^{11}$ $1.6\times10^9$ $2.3\times10^3$ | | Wang et al. (2017) Wang et al. (2017) Wang et al. (2017) | Q Q Q | 80, 238 80, 239 80, 240 |
| MCM:IPBZMUCCO $C_9H_{12}O_5$ XOMPDNKHXMMXOK-UHFFFAOYSA-N | $3.2\times10^8$ $1.3\times10^7$ $5.4\times10^2$ | | Wang et al. (2017) Wang et al. (2017) Wang et al. (2017) | Q Q Q | 80, 238 80, 239 80, 240 |
| MCM:IPBZMUCOH $C_9H_{14}O_5$ AKVRVUIGBHJRNQ-UHFFFAOYSA-N | $5.3\times10^8$ $6.5\times10^8$ $2.2\times10^4$ | | Wang et al. (2017) Wang et al. (2017) Wang et al. (2017) | Q Q Q | 80, 238 80, 239 80, 240 |
| MCM:IPBZMUCOOH $C_9H_{14}O_6$ LSOCHQKLDSXALN-UHFFFAOYSA-N | $8.5\times10^{11}$ $2.1\times10^9$ $1.0\times10^4$ | | Wang et al. (2017) Wang et al. (2017) Wang et al. (2017) | Q Q Q | 80, 238 80, 239 80, 240 |
| MCM:METLMUCCO $C_9H_{12}O_5$ ZRMCSOCJQIXUEX-UHFFFAOYSA-N | $1.9\times10^8$ $3.7\times10^6$ $5.8\times10^2$ | | Wang et al. (2017) Wang et al. (2017) Wang et al. (2017) | Q Q Q | 80, 238 80, 239 80, 240 |
| MCM:METLMUCOH $C_9H_{14}O_5$ ARJLZFCPXRPKBS-UHFFFAOYSA-N | $3.0\times10^8$ $1.8\times10^8$ $1.6\times10^4$ | | Wang et al. (2017) Wang et al. (2017) Wang et al. (2017) | Q Q Q | 80, 238 80, 239 80, 240 |
| MCM:METLMUCOOH $C_9H_{14}O_6$ UXABECSPQFCGIJ-UHFFFAOYSA-N | $5.0\times10^{11}$ $6.0\times10^8$ $2.3\times10^4$ | | Wang et al. (2017) Wang et al. (2017) Wang et al. (2017) | Q Q Q | 80, 238 80, 239 80, 240 |





Table A3.9: Ethers (ROR) (. . . continued)

| Substance Formula (Trivial Name) [CAS Registry Number] InChIKey | $H_s^{cp}$ (at $T^\ominus$) $\left[\dfrac{\text{mol}}{\text{m}^3\,\text{Pa}}\right]$ | $\dfrac{\text{d}\ln H_s^{cp}}{\text{d}(1/T)}$ [K] | Reference | Type | Note |
|---|---|---|---|---|---|
| MCM:OETLMUCCO | $1.6\times10^8$ | | Wang et al. (2017) | Q | 80, 238 |
| $C_9H_{12}O_5$ | $2.0\times10^6$ | | Wang et al. (2017) | Q | 80, 239 |
| JSKKVRLLVVSVCZ-UHFFFAOYSA-N | $9.6\times10^1$ | | Wang et al. (2017) | Q | 80, 240 |
| MCM:PBZMUCCO | $2.7\times10^8$ | | Wang et al. (2017) | Q | 80, 238 |
| $C_9H_{12}O_5$ | $1.1\times10^7$ | | Wang et al. (2017) | Q | 80, 239 |
| FCRYPGHUONBQCX-UHFFFAOYSA-N | $9.1\times10^2$ | | Wang et al. (2017) | Q | 80, 240 |
| MCM:PBZMUCOH | $4.4\times10^8$ | | Wang et al. (2017) | Q | 80, 238 |
| $C_9H_{14}O_5$ | $5.8\times10^8$ | | Wang et al. (2017) | Q | 80, 239 |
| VYXNVIKZUYYSSA-UHFFFAOYSA-N | $1.9\times10^4$ | | Wang et al. (2017) | Q | 80, 240 |
| MCM:PBZMUCOOH | $7.3\times10^{11}$ | | Wang et al. (2017) | Q | 80, 238 |
| $C_9H_{14}O_6$ | $2.0\times10^9$ | | Wang et al. (2017) | Q | 80, 239 |
| FCKGMMNUJIAHAV-UHFFFAOYSA-N | $5.0\times10^4$ | | Wang et al. (2017) | Q | 80, 240 |
| MCM:PETLMUCCO | $1.9\times10^8$ | | Wang et al. (2017) | Q | 80, 238 |
| $C_9H_{12}O_5$ | $5.9\times10^6$ | | Wang et al. (2017) | Q | 80, 239 |
| RORTYMWKJCCJQU-UHFFFAOYSA-N | $6.8\times10^2$ | | Wang et al. (2017) | Q | 80, 240 |
| MCM:PETLMUCOH | $3.0\times10^8$ | | Wang et al. (2017) | Q | 80, 238 |
| $C_9H_{14}O_5$ | $2.5\times10^8$ | | Wang et al. (2017) | Q | 80, 239 |
| ZLOWXLKYIJYQIH-UHFFFAOYSA-N | $8.7\times10^3$ | | Wang et al. (2017) | Q | 80, 240 |
| MCM:PETLMUCOOH | $5.0\times10^{11}$ | | Wang et al. (2017) | Q | 80, 238 |
| $C_9H_{14}O_6$ | $8.1\times10^8$ | | Wang et al. (2017) | Q | 80, 239 |
| RSNULGXAYPPPGL-UHFFFAOYSA-N | $3.3\times10^4$ | | Wang et al. (2017) | Q | 80, 240 |
| MCM:TM124MUCCO | $2.1\times10^7$ | | Wang et al. (2017) | Q | 80, 238 |
| $C_9H_{12}O_5$ | $9.8\times10^6$ | | Wang et al. (2017) | Q | 80, 239 |
| DQECTOJLTKBSTJ-UHFFFAOYSA-N | $1.5\times10^3$ | | Wang et al. (2017) | Q | 80, 240 |
| MCM:TM124MUCOH | $2.1\times10^8$ | | Wang et al. (2017) | Q | 80, 238 |
| $C_9H_{14}O_5$ | $1.0\times10^8$ | | Wang et al. (2017) | Q | 80, 239 |
| KPYCPCVRGFCQPX-UHFFFAOYSA-N | $7.6\times10^3$ | | Wang et al. (2017) | Q | 80, 240 |
| MCM:TM124MUOOH | $1.4\times10^{10}$ | | Wang et al. (2017) | Q | 80, 238 |
| $C_9H_{14}O_6$ | $6.8\times10^7$ | | Wang et al. (2017) | Q | 80, 239 |
| DQFPDDPNMNSRMG-UHFFFAOYSA-N | $3.5\times10^3$ | | Wang et al. (2017) | Q | 80, 240 |
| MCM:TM135MUOH | $2.1\times10^8$ | | Wang et al. (2017) | Q | 80, 238 |
| $C_9H_{14}O_5$ | $2.4\times10^8$ | | Wang et al. (2017) | Q | 80, 239 |
| CZQCDAJVGDGQTR-UHFFFAOYSA-N | $1.4\times10^4$ | | Wang et al. (2017) | Q | 80, 240 |
| MCM:TM135MUOOH | $1.4\times10^{10}$ | | Wang et al. (2017) | Q | 80, 238 |
| $C_9H_{14}O_6$ | $1.2\times10^8$ | | Wang et al. (2017) | Q | 80, 239 |
| FIFCCDBMOKNVLN-UHFFFAOYSA-N | $2.6\times10^3$ | | Wang et al. (2017) | Q | 80, 240 |
| MCM:DMEBMUOH | $1.7\times10^8$ | | Wang et al. (2017) | Q | 80, 238 |
| $C_{10}H_{16}O_5$ | $1.3\times10^8$ | | Wang et al. (2017) | Q | 80, 239 |
| MURMIEBRJYKLPQ-UHFFFAOYSA-N | $1.1\times10^4$ | | Wang et al. (2017) | Q | 80, 240 |



Table A3.9: Ethers (ROR) (...continued)

| Substance Formula (Trivial Name) [CAS Registry Number] InChIKey | $H_s^{cp}$ (at $T^\ominus$) $\left[\dfrac{\text{mol}}{\text{m}^3\,\text{Pa}}\right]$ | $\dfrac{\text{d}\ln H_s^{cp}}{\text{d}(1/T)}$ [K] | Reference | Type | Note |
|---|---|---|---|---|---|
| MCM:DMEBMUOOH $C_{10}H_{16}O_6$ QHUZCFCKANZAOE-UHFFFAOYSA-N | $1.2\times10^{10}$ $6.2\times10^{7}$ $3.7\times10^{2}$ | | Wang et al. (2017) Wang et al. (2017) Wang et al. (2017) | Q Q Q | 80, 238 80, 239 80, 240 |
| MCM:DETLMUOH $C_{11}H_{18}O_5$ BLCXZNPOSIAMRZ-UHFFFAOYSA-N | $1.4\times10^{8}$ $8.5\times10^{7}$ $5.4\times10^{3}$ | | Wang et al. (2017) Wang et al. (2017) Wang et al. (2017) | Q Q Q | 80, 238 80, 239 80, 240 |
| MCM:DETLMUOOH $C_{11}H_{18}O_6$ ZGRWZBBDFVAIEE-UHFFFAOYSA-N | $9.8\times10^{9}$ $4.2\times10^{7}$ $1.4\times10^{2}$ | | Wang et al. (2017) Wang et al. (2017) Wang et al. (2017) | Q Q Q | 80, 238 80, 239 80, 240 |
| MCM:ETOMECO2H $C_4H_8O_3$ YZGQDNOIGFBYKF-UHFFFAOYSA-N | $2.8\times10^{2}$ $3.6\times10^{2}$ $7.6\times10^{2}$ | | Wang et al. (2017) Wang et al. (2017) Wang et al. (2017) | Q Q Q | 80, 238 80, 239 80, 240 |
| MCM:MTBEBCO2H $C_5H_{10}O_3$ BKBZFJRHYSCZQA-UHFFFAOYSA-N | $1.6\times10^{2}$ $3.9\times10^{1}$ $8.7\times10^{1}$ | | Wang et al. (2017) Wang et al. (2017) Wang et al. (2017) | Q Q Q | 80, 238 80, 239 80, 240 |
| MCM:BOXMCO2H $C_6H_{12}O_3$ AJQOASGWDCBKCJ-UHFFFAOYSA-N | $1.8\times10^{2}$ $1.2\times10^{2}$ $1.6\times10^{2}$ | | Wang et al. (2017) Wang et al. (2017) Wang et al. (2017) | Q Q Q | 80, 238 80, 239 80, 240 |
| MCM:EPXDLCO2H $C_4H_4O_4$ BIFCJMYQXQSHFE-UHFFFAOYSA-N | $9.3\times10^{5}$ $1.5\times10^{5}$ $5.4\times10^{4}$ | | Wang et al. (2017) Wang et al. (2017) Wang et al. (2017) | Q Q Q | 80, 238 80, 239 80, 240 |
| MCM:EPXMDLCO2H $C_5H_6O_4$ HNJRQDPCPFMZIJ-UHFFFAOYSA-N | $5.4\times10^{5}$ $5.9\times10^{4}$ $2.7\times10^{4}$ | | Wang et al. (2017) Wang et al. (2017) Wang et al. (2017) | Q Q Q | 80, 238 80, 239 80, 240 |
| MCM:BZEMUCCO2H $C_6H_6O_4$ YYDFWEKDUXDLMC-UHFFFAOYSA-N | $2.5\times10^{6}$ $4.8\times10^{5}$ $3.7\times10^{6}$ | | Wang et al. (2017) Wang et al. (2017) Wang et al. (2017) | Q Q Q | 80, 238 80, 239 80, 240 |
| MCM:EPXM2DCO2H $C_6H_8O_4$ TVFMSHFDVSCCSQ-UHFFFAOYSA-N | $2.9\times10^{5}$ $2.6\times10^{4}$ $6.8\times10^{3}$ | | Wang et al. (2017) Wang et al. (2017) Wang et al. (2017) | Q Q Q | 80, 238 80, 239 80, 240 |
| MCM:EPXMEDCO2H $C_7H_{10}O_4$ CTYOPZSWGBYACX-UHFFFAOYSA-N | $2.3\times10^{5}$ $1.5\times10^{4}$ $5.1\times10^{3}$ | | Wang et al. (2017) Wang et al. (2017) Wang et al. (2017) | Q Q Q | 80, 238 80, 239 80, 240 |
| MCM:OXYMUCCO2H $C_8H_{10}O_4$ CDVXJWVOWNWWPU-UHFFFAOYSA-N | $7.6\times10^{5}$ $8.1\times10^{4}$ $7.1\times10^{5}$ | | Wang et al. (2017) Wang et al. (2017) Wang et al. (2017) | Q Q Q | 80, 238 80, 239 80, 240 |
| MCM:OETLMUCO2H $C_9H_{12}O_4$ UVQOKFRFLASUOJ-UHFFFAOYSA-N | $6.8\times10^{5}$ $5.1\times10^{4}$ $3.1\times10^{5}$ | | Wang et al. (2017) Wang et al. (2017) Wang et al. (2017) | Q Q Q | 80, 238 80, 239 80, 240 |



Table A3.9: Ethers (ROR) (. . . continued)

| Substance<br>Formula<br>(Trivial Name)<br>[CAS Registry Number]<br>InChIKey | $H_s^{cp}$<br>(at $T^\ominus$)<br>$\left[\dfrac{\mathrm{mol}}{\mathrm{m}^3\,\mathrm{Pa}}\right]$ | $\dfrac{\mathrm{d}\ln H_s^{cp}}{\mathrm{d}(1/T)}$<br><br>[K] | Reference | Type | Note |
|---|---|---|---|---|---|
| MCM:EPXKTMCO2H<br>$C_6H_8O_4$<br>BXVGKWFSHJVVPB-UHFFFAOYSA-N | $3.1\times10^5$<br>$5.3\times10^4$<br>$6.6\times10^4$ | | Wang et al. (2017)<br>Wang et al. (2017)<br>Wang et al. (2017) | Q<br>Q<br>Q | 80, 238<br>80, 239<br>80, 240 |
| MCM:EPXMKTCO2H<br>$C_6H_8O_4$<br>FTLSPZDBESPPRR-UHFFFAOYSA-N | $3.1\times10^5$<br>$1.1\times10^5$<br>$3.3\times10^4$ | | Wang et al. (2017)<br>Wang et al. (2017)<br>Wang et al. (2017) | Q<br>Q<br>Q | 80, 238<br>80, 239<br>80, 240 |
| MCM:EPXEKTCO2H<br>$C_7H_{10}O_4$<br>XBMXXNQRKRSRRJ-UHFFFAOYSA-N | $2.8\times10^5$<br>$1.8\times10^5$<br>$5.0\times10^5$ | | Wang et al. (2017)<br>Wang et al. (2017)<br>Wang et al. (2017) | Q<br>Q<br>Q | 80, 238<br>80, 239<br>80, 240 |
| MCM:TLEMUCCO2H<br>$C_7H_8O_4$<br>YZQFSRDEODCLRC-UHFFFAOYSA-N | $1.7\times10^6$<br>$8.3\times10^5$<br>$1.3\times10^7$ | | Wang et al. (2017)<br>Wang et al. (2017)<br>Wang et al. (2017) | Q<br>Q<br>Q | 80, 238<br>80, 239<br>80, 240 |
| MCM:EBZMUCCO2H<br>$C_8H_{10}O_4$<br>BSMJHSATRRVHTI-UHFFFAOYSA-N | $1.5\times10^6$<br>$5.0\times10^5$<br>$6.8\times10^6$ | | Wang et al. (2017)<br>Wang et al. (2017)<br>Wang et al. (2017) | Q<br>Q<br>Q | 80, 238<br>80, 239<br>80, 240 |
| MCM:MXYMUCCO2H<br>$C_8H_{10}O_4$<br>BMJHPUYAWBLOER-UHFFFAOYSA-N | $9.1\times10^5$<br>$4.2\times10^5$<br>$3.2\times10^6$ | | Wang et al. (2017)<br>Wang et al. (2017)<br>Wang et al. (2017) | Q<br>Q<br>Q | 80, 238<br>80, 239<br>80, 240 |
| MCM:PXYMUCCO2H<br>$C_8H_{10}O_4$<br>FMBYVAMRNZYNGO-UHFFFAOYSA-N | $9.1\times10^5$<br>$2.9\times10^5$<br>$8.3\times10^6$ | | Wang et al. (2017)<br>Wang et al. (2017)<br>Wang et al. (2017) | Q<br>Q<br>Q | 80, 238<br>80, 239<br>80, 240 |
| MCM:IPBZMUCO2H<br>$C_9H_{12}O_4$<br>LXYULRJMRUJREP-UHFFFAOYSA-N | $1.4\times10^6$<br>$3.2\times10^5$<br>$2.2\times10^6$ | | Wang et al. (2017)<br>Wang et al. (2017)<br>Wang et al. (2017) | Q<br>Q<br>Q | 80, 238<br>80, 239<br>80, 240 |
| MCM:METLMUCO2H<br>$C_9H_{12}O_4$<br>TZAADSLPRMCSBM-UHFFFAOYSA-N | $8.0\times10^5$<br>$2.1\times10^5$<br>$1.0\times10^6$ | | Wang et al. (2017)<br>Wang et al. (2017)<br>Wang et al. (2017) | Q<br>Q<br>Q | 80, 238<br>80, 239<br>80, 240 |
| MCM:PBZMUCCO2H<br>$C_9H_{12}O_4$<br>MSVJIRWCUUEXSY-UHFFFAOYSA-N | $1.2\times10^6$<br>$3.0\times10^5$<br>$2.9\times10^6$ | | Wang et al. (2017)<br>Wang et al. (2017)<br>Wang et al. (2017) | Q<br>Q<br>Q | 80, 238<br>80, 239<br>80, 240 |
| MCM:PETLMUCO2H<br>$C_9H_{12}O_4$<br>YZGPSCFZKHZHEQ-UHFFFAOYSA-N | $8.0\times10^5$<br>$1.4\times10^5$<br>$2.6\times10^6$ | | Wang et al. (2017)<br>Wang et al. (2017)<br>Wang et al. (2017) | Q<br>Q<br>Q | 80, 238<br>80, 239<br>80, 240 |
| MCM:TM124MUO2H<br>$C_9H_{12}O_4$<br>VDBMALGYQLXWML-UHFFFAOYSA-N | $4.9\times10^5$<br>$1.4\times10^5$<br>$4.3\times10^6$ | | Wang et al. (2017)<br>Wang et al. (2017)<br>Wang et al. (2017) | Q<br>Q<br>Q | 80, 238<br>80, 239<br>80, 240 |
| MCM:TM135MUO2H<br>$C_9H_{12}O_4$<br>RLMHTEATULORAU-UHFFFAOYSA-N | $6.2\times10^5$<br>$3.8\times10^5$<br>$2.1\times10^6$ | | Wang et al. (2017)<br>Wang et al. (2017)<br>Wang et al. (2017) | Q<br>Q<br>Q | 80, 238<br>80, 239<br>80, 240 |





Table A3.9: Ethers (ROR) (...continued)

| Substance Formula (Trivial Name) [CAS Registry Number] InChIKey | $H_s^{cp}$ (at $T^{\ominus}$) $\left[\dfrac{\text{mol}}{\text{m}^3\,\text{Pa}}\right]$ | $\dfrac{\text{d}\ln H_s^{cp}}{\text{d}(1/T)}$ [K] | Reference | Type | Note |
|---|---|---|---|---|---|
| MCM:DMEBMUO2H $C_{10}H_{14}O_4$ GTDCDBIZJZUTJP-UHFFFAOYSA-N | $5.4\times10^5$ $2.0\times10^5$ $1.0\times10^6$ | | Wang et al. (2017) Wang et al. (2017) Wang et al. (2017) | Q Q Q | 80, 238 80, 239 80, 240 |
| MCM:DETLMUO2H $C_{11}H_{16}O_4$ FTAMRHAZJZCWNK-UHFFFAOYSA-N | $4.5\times10^5$ $1.4\times10^5$ $5.5\times10^5$ | | Wang et al. (2017) Wang et al. (2017) Wang et al. (2017) | Q Q Q | 80, 238 80, 239 80, 240 |
| MCM:DMCOOH $C_3H_6O_5$ ISFGCMQHKVEPTP-UHFFFAOYSA-N | $2.0\times10^4$ $3.7\times10^3$ $1.9\times10^2$ | | Wang et al. (2017) Wang et al. (2017) Wang et al. (2017) | Q Q Q | 80, 238 80, 239 80, 240 |
| MCM:MMF $C_3H_6O_3$ VKWJMTLAAJULGF-UHFFFAOYSA-N | $1.2$ $2.1$ $1.5$ | | Wang et al. (2017) Wang et al. (2017) Wang et al. (2017) | Q Q Q | 80, 238 80, 239 80, 240 |
| MCM:MMFOOH $C_3H_6O_5$ AAAZTRZXKHTMLL-UHFFFAOYSA-N | $9.8\times10^4$ $5.6\times10^2$ $9.6\times10^1$ | | Wang et al. (2017) Wang et al. (2017) Wang et al. (2017) | Q Q Q | 80, 238 80, 239 80, 240 |
| MCM:DMCOH $C_3H_6O_4$ YGODNDLTUXBUQH-UHFFFAOYSA-N | $7.4\times10^2$ $7.4\times10^2$ $6.9\times10^1$ | | Wang et al. (2017) Wang et al. (2017) Wang et al. (2017) | Q Q Q | 80, 238 80, 239 80, 240 |
| MCM:MMFOH $C_3H_6O_4$ GDDJZBRTXIGGHV-UHFFFAOYSA-N | $3.6\times10^3$ $1.5\times10^2$ $4.5\times10^1$ | | Wang et al. (2017) Wang et al. (2017) Wang et al. (2017) | Q Q Q | 80, 238 80, 239 80, 240 |
| GOLIG1 $C_4H_6O_5$ LEKXYOUPWKVTGM-UHFFFAOYSA-N | $1.5\times10^{12}$ | 19000 | Wieser et al. (2023) | Q | 437 |
| GOLIG2 $C_4H_8O_6$ GWSRJJGBNXJVPO-UHFFFAOYSA-N | $1.5\times10^{15}$ | 24000 | Wieser et al. (2023) | Q | 437 |
| GOLIG3 $C_4H_{10}O_7$ ADECTKUVLJSMDK-UHFFFAOYSA-N | $3.8\times10^{18}$ | 29000 | Wieser et al. (2023) | Q | 437 |
| MGLYFB $C_6H_{10}O_6$ UCPBJDKPOQEXSQ-UHFFFAOYSA-N | $7.1\times10^8$ | 23000 | Wieser et al. (2023) | Q | 437 |
| MGLYOXDA $C_6H_{10}O_5$ QMSLRDMQAWWKIZ-UHFFFAOYSA-N | $2.0\times10^8$ | 16000 | Wieser et al. (2023) | Q | 437 |
| MGLYOXDB $C_6H_{12}O_6$ IUHMFBWAUOLAFM-UHFFFAOYSA-N | $1.5\times10^{13}$ | 22000 | Wieser et al. (2023) | Q | 437 |





### A3.10 Heterocycles with oxygen

Table A3.10: Heterocycles with oxygen

| Substance<br>Formula<br>(Trivial Name)<br>[CAS Registry Number]<br>InChIKey | $H_s^{cp}$<br>(at $T^\ominus$)<br>$\left[\dfrac{\mathrm{mol}}{\mathrm{m^3\,Pa}}\right]$ | $\dfrac{\mathrm{d}\ln H_s^{cp}}{\mathrm{d}(1/T)}$<br><br>[K] | Reference | Type | Note |
|---|---|---|---|---|---|
| $\beta$-propiolactone<br>$C_3H_4O_2$<br>[57-57-8]<br>VEZXCJBBBCKRPI-UHFFFAOYSA-N | $1.3\times10^1$ | | Ebert et al. (2023) | ? | 316 |
| 2-methyl-1,3-dioxolane<br>$C_4H_8O_2$<br>[497-26-7]<br>HTWIZMNMTWYQRN-UHFFFAOYSA-N | $7.9\times10^{-1}$ | | Ebert et al. (2023) | ? | 316 |
| DIEPOXO3<br>$C_5H_8O_3$<br>QGQUFAZKBSHWQB-UHFFFAOYSA-N | $1.3\times10^3$ | 14000 | Wieser et al. (2023) | Q | 437 |
| IEPOXO4<br>$C_5H_8O_3$<br>DNQIATHBSKHPLN-UHFFFAOYSA-N | $2.1\times10^4$ | 14000 | Wieser et al. (2023) | Q | 437 |
| 2,2-dimethyl-1,3-dioxolane<br>$C_5H_{10}O_2$<br>[2916-31-6]<br>SIJBDWPVNAYVGY-UHFFFAOYSA-N | $4.8\times10^{-1}$ | | Ebert et al. (2023) | ? | 316 |
| METHFDIOL<br>$C_5H_{10}O_3$<br>UNAIOYXCADSMDR-UHFFFAOYSA-N | $9.0\times10^5$ | 11000 | Wieser et al. (2023) | Q | 437 |
| 3,4-dihydro-2-methoxy-2H-pyran<br>$C_6H_{10}O_2$<br>[4454-05-1]<br>XCYWUZHUTJDTGS-UHFFFAOYSA-N | $9.7\times10^{-2}$ | | Ebert et al. (2023) | ? | 316 |
| maltol<br>$C_6H_6O_3$<br>[118-71-8]<br>XPCTZQVDEJYUGT-UHFFFAOYSA-N | $7.3\times10^2$ | | Abraham et al. (2019) | Q | |
| 2-methylfuran<br>$C_5H_6O$<br>[534-22-5]<br>VQKFNUFAXTZWDK-UHFFFAOYSA-N | $1.5\times10^{-3}$ | 4100 | Wieland et al. (2015) | M | 532 |
| 3-methylfuran<br>$C_5H_6O$<br>[930-27-8]<br>KJRRQXYWFQKJIP-UHFFFAOYSA-N | $3.2\times10^{-3}$ | | Wu et al. (2022a) | Q | 413 |





Table A3.10: Heterocycles with oxygen (. . . continued)

| Substance Formula (Trivial Name) [CAS Registry Number] InChIKey | $H_s^{cp}$ (at $T^{\ominus}$) $\left[\dfrac{\text{mol}}{\text{m}^3\,\text{Pa}}\right]$ | $\dfrac{\text{d}\ln H_s^{cp}}{\text{d}(1/T)}$ [K] | Reference | Type | Note |
|---|---|---|---|---|---|
| 2-furanmethanol | $1.3\times10^2$ | | Duchowicz et al. (2020) | V | 186 |
| $C_5H_6O_2$ | $1.2\times10^2$ | | HSDB (2015) | V | |
| (furfuryl alcohol) | $2.5\times10^1$ | | Yaws (2003) | X | 258 |
| [98-00-0] | $2.2\times10^1$ | | Dupeux et al. (2022) | Q | 259 |
| XPFVYQJUAUNWIW-UHFFFAOYSA-N | $2.4\times10^1$ | | Duchowicz et al. (2020) | Q | |
| | $5.3\times10^1$ | | Gharagheizi et al. (2012) | Q | |
| | $6.2\times10^1$ | | Raventos-Duran et al. (2010) | Q | 271, 243 |
| | $3.9\times10^1$ | | Raventos-Duran et al. (2010) | Q | 244 |
| | $4.9\times10^1$ | | Raventos-Duran et al. (2010) | Q | 245 |
| | $3.4\times10^1$ | | Hilal et al. (2008) | Q | |
| | $4.8\times10^1$ | | Modarresi et al. (2007) | Q | 67 |
| | $4.8\times10^2$ | | Katritzky et al. (1998) | Q | |
| | $1.1\times10^2$ | | Yaws (1999) | ? | 21, 12 |
| tetrahydropyran-2-methanol | $6.5\times10^2$ | | Duchowicz et al. (2020) | V | 186 |
| $C_6H_{12}O_2$ | $5.2\times10^1$ | | Duchowicz et al. (2020) | Q | |
| [100-72-1] | $6.2\times10^1$ | | Raventos-Duran et al. (2010) | Q | 271, 243 |
| ROTONRWJLXYJBD-UHFFFAOYSA-N | $3.1\times10^2$ | | Raventos-Duran et al. (2010) | Q | 244 |
| | $2.0\times10^3$ | | Raventos-Duran et al. (2010) | Q | 245 |
| | $9.0\times10^1$ | | Hilal et al. (2008) | Q | |
| | $8.5\times10^1$ | | Modarresi et al. (2007) | Q | 67 |
| oxirane | $5.8\times10^{-2}$ | 3200 | Conway et al. (1983) | M | |
| $C_2H_4O$ | $8.3\times10^{-2}$ | | Lide and Frederikse (1995) | V | |
| (ethylene oxide) | $8.6\times10^{-2}$ | | Mackay et al. (1993) | V | |
| [75-21-8] | $5.0\times10^{-2}$ | | Hwang et al. (1992) | V | |
| IAYPIBMASNFSPL-UHFFFAOYSA-N | $6.0\times10^{-2}$ | | Keshavarz et al. (2022) | Q | |
| | $1.4\times10^{-1}$ | | Duchowicz et al. (2020) | Q | |
| | $9.3\times10^{-3}$ | | Wang et al. (2017) | Q | 80, 238 |
| | $5.9\times10^{-2}$ | | Wang et al. (2017) | Q | 80, 239 |
| | $6.0\times10^{-2}$ | | Wang et al. (2017) | Q | 80, 240 |
| | $3.9\times10^{-2}$ | | Hilal et al. (2008) | Q | |
| | $2.9\times10^{-2}$ | | Modarresi et al. (2007) | Q | 67 |
| | $6.7\times10^{-2}$ | | Yaffe et al. (2003) | Q | 248, 249 |
| | $1.0\times10^{-2}$ | | Katritzky et al. (1998) | Q | |
| | $6.7\times10^{-2}$ | | Duchowicz et al. (2020) | ? | 185, 21 |
| | $4.1\times10^{-2}$ | | Yaws (1999) | ? | 21 |
| 1,2-epoxypropane | $1.4\times10^{-1}$ | | Duchowicz et al. (2020) | V | 186 |
| $C_3H_6O$ | $1.4\times10^{-1}$ | | HSDB (2015) | V | |
| (1,2-propylene oxide) | $1.2\times10^{-1}$ | | Mackay et al. (2006c) | V | |
| [75-56-9] | $1.2\times10^{-1}$ | | Lide and Frederikse (1995) | V | |
| GOOHAUXETOMSMM-UHFFFAOYSA-N | $1.2\times10^{-1}$ | | Mackay et al. (1993) | V | |
| | $1.7\times10^{-1}$ | | Yaws (2003) | X | 237, 12 |
| | $5.2\times10^{-2}$ | | Goldstein (1982) | X | 446 |
| | $5.1\times10^{-2}$ | 3500 | Goldstein (1982) | X | 298 |
| | $6.3\times10^{-2}$ | | Duchowicz et al. (2020) | Q | |
| | $1.2\times10^{-1}$ | | Gharagheizi et al. (2010) | Q | 246 |
| | $1.7\times10^{-2}$ | | Hilal et al. (2008) | Q | |



Table A3.10: Heterocycles with oxygen (...continued)

| Substance Formula (Trivial Name) [CAS Registry Number] InChIKey | $H_s^{cp}$ (at $T^\ominus$) $\left[\dfrac{\text{mol}}{\text{m}^3\,\text{Pa}}\right]$ | $\dfrac{\text{d}\ln H_s^{cp}}{\text{d}(1/T)}$ [K] | Reference | Type | Note |
|---|---|---|---|---|---|
| | $1.5\times10^{-1}$ | | Yaffe et al. (2003) | Q | 248, 249 |
| | $1.3\times10^{-2}$ | | Katritzky et al. (1998) | Q | |
| | $1.4\times10^{-1}$ | | Yaws (1999) | ? | 21, 12 |
| phenyloxirane | $6.2\times10^{-1}$ | | HSDB (2015) | V | |
| $C_8H_8O$ | $5.8\times10^{-1}$ | | Mackay et al. (2006c) | V | |
| (styrene oxide) | $5.8\times10^{-1}$ | | Mackay et al. (1993) | V | |
| [96-09-3] | $6.2\times10^{-1}$ | | Meylan and Howard (1991) | V | |
| AWMVMTVKBNGEAK-UHFFFAOYSA-N | $2.5\times10^{-1}$ | | Hilal et al. (2008) | Q | |
| | $5.5\times10^{-1}$ | | Modarresi et al. (2007) | Q | 67 |
| | 1.0 | | Meylan and Howard (1991) | Q | |
| oxacyclopentadiene | $1.8\times10^{-3}$ | | HSDB (2015) | V | |
| $C_4H_4O$ | $1.8\times10^{-3}$ | | Mackay et al. (2006c) | V | |
| (furan; furfuran) | $1.8\times10^{-3}$ | | Mackay et al. (1993) | V | |
| [110-00-9] | $1.8\times10^{-3}$ | | Yaws (2003) | X | 258 |
| YLQBMQCUIZJEEH-UHFFFAOYSA-N | $1.8\times10^{-3}$ | | Yaws (2003) | X | 237 |
| | $2.2\times10^{-3}$ | | Dupeux et al. (2022) | Q | 259 |
| | $4.6\times10^{-3}$ | | Hayer et al. (2022) | Q | 20 |
| | $3.1\times10^{-2}$ | | Raventos-Duran et al. (2010) | Q | 242, 243 |
| | $2.0\times10^{-3}$ | | Raventos-Duran et al. (2010) | Q | 244 |
| | $2.0\times10^{-3}$ | | Raventos-Duran et al. (2010) | Q | 245 |
| | $1.9\times10^{-3}$ | | Gharagheizi et al. (2010) | Q | 246 |
| | $2.3\times10^{-3}$ | | Hilal et al. (2008) | Q | |
| | $1.9\times10^{-3}$ | | Yaffe et al. (2003) | Q | 248, 249 |
| | $4.3\times10^{-3}$ | | Yao et al. (2002) | Q | 229 |
| | $1.2\times10^{-2}$ | | Katritzky et al. (1998) | Q | |
| | $1.8\times10^{-3}$ | | Yaws (1999) | ? | 21 |
| | $1.8\times10^{-3}$ | | Yaws and Yang (1992) | ? | 21 |
| dibenzofuran | $7.4\times10^{-2}$ | 5800 | Brockbank (2013) | L | 1 |
| $C_{12}H_8O$ | $4.6\times10^{-2}$ | | Duchowicz et al. (2020) | V | 186 |
| (2,2'-biphenylene oxide) | $4.7\times10^{-2}$ | | HSDB (2015) | V | |
| [132-64-9] | $7.1\times10^{-2}$ | | Mackay et al. (2006b) | V | |
| TXCDCPKCNAJMEE-UHFFFAOYSA-N | $7.2\times10^{-2}$ | | Govers and Krop (1998) | V | |
| | $9.1\times10^{-2}$ | | Mackay et al. (1992b) | X | 364 |
| | $8.9\times10^{-3}$ | | Yaws (2003) | X | 237 |
| | $2.8\times10^{-1}$ | | Duchowicz et al. (2020) | Q | |
| | $9.0\times10^{-2}$ | | Gharagheizi et al. (2010) | Q | 246 |
| | $8.2\times10^{-2}$ | | Saçan et al. (2005) | Q | |
| | $4.7\times10^{-2}$ | | Govers and Krop (1998) | Q | |
| 2-furancarboxaldehyde | 2.6 | | Duchowicz et al. (2020) | V | 186 |
| $C_5H_4O_2$ | 2.6 | | HSDB (2015) | V | |
| (furfural; 2-furanaldehyde) | 2.7 | | Mackay et al. (2006c) | V | |
| [98-01-1] | 2.7 | | Mackay et al. (1995) | V | |
| HYBBIBNJHNGZAN-UHFFFAOYSA-N | 3.0 | | Yaws (2003) | X | 258 |
| | 3.0 | | Yaws (2003) | X | 237 |
| | 9.6 | | Dupeux et al. (2022) | Q | 259 |





Table A3.10: Heterocycles with oxygen (...continued)

| Substance Formula (Trivial Name) [CAS Registry Number] InChIKey | $H_s^{cp}$ (at $T^{\ominus}$) $\left[\dfrac{\mathrm{mol}}{\mathrm{m^3\,Pa}}\right]$ | $\dfrac{\mathrm{d}\ln H_s^{cp}}{\mathrm{d}(1/T)}$ [K] | Reference | Type | Note |
|---|---|---|---|---|---|
| | $8.4\times10^{-1}$ | | Duchowicz et al. (2020) | Q | |
| | 1.7 | | Gharagheizi et al. (2012) | Q | |
| | $2.0\times10^{-1}$ | | Raventos-Duran et al. (2010) | Q | 271, 243 |
| | 6.2 | | Raventos-Duran et al. (2010) | Q | 244 |
| | $7.8\times10^{-1}$ | | Raventos-Duran et al. (2010) | Q | 245 |
| | 3.0 | | Gharagheizi et al. (2010) | Q | 246 |
| | 6.0 | | Hilal et al. (2008) | Q | |
| | $8.1\times10^{-1}$ | | Modarresi et al. (2007) | Q | 67 |
| | $7.2\times10^{-2}$ | | Emel'yanenko et al. (2007) | Q | 415 |
| | $7.2\times10^{-2}$ | | Hertel and Sommer (2006) | Q | 415 |
| | | 6100 | Kühne et al. (2005) | Q | |
| | | 5900 | Kühne et al. (2005) | ? | |
| | 3.0 | | Yaws (1999) | ? | 21 |
| 5-methylfurfural $C_6H_6O_2$ [620-02-0] OUDFNZMQXZILJD-UHFFFAOYSA-N | 4.7 | | Ebert et al. (2023) | ? | 316 |
| 2,2-dimethyloxirane $C_4H_8O$ (isobutylene oxide) [558-30-5] GELKGHVAFRCJNA-UHFFFAOYSA-N | $2.9\times10^{-2}$ | | Ebert et al. (2023) | ? | 316 |
| tetrahydrofuran $C_4H_8O$ (THF) [109-99-9] WYURNTSHIVDZCO-UHFFFAOYSA-N | $1.5\times10^{-1}$ | 5700 | Brockbank (2013) | L | 1 |
| | $1.5\times10^{-1}$ | 5700 | Ondo and Dohnal (2007) | M | 1 |
| | $6.4\times10^{-1}$ | | Welke et al. (1998) | M | |
| | $6.0\times10^{-1}$ | 4200 | Pividal et al. (1992) | M | |
| | $2.2\times10^{-1}$ | | Signer et al. (1969) | M | |
| | $1.4\times10^{-1}$ | 5700 | Cabani et al. (1971b) | T | |
| | $6.0\times10^{-2}$ | | Yaws (2003) | X | 237, 12 |
| | $1.1\times10^{-1}$ | | Keshavarz et al. (2022) | Q | |
| | $1.3\times10^{-1}$ | | Duchowicz et al. (2020) | Q | |
| | $8.5\times10^{-2}$ | | Gharagheizi et al. (2012) | Q | |
| | $2.5\times10^{-2}$ | | Raventos-Duran et al. (2010) | Q | 242, 243 |
| | $2.0\times10^{-1}$ | | Raventos-Duran et al. (2010) | Q | 244 |
| | $1.2\times10^{-1}$ | | Raventos-Duran et al. (2010) | Q | 245 |
| | $6.0\times10^{-2}$ | | Gharagheizi et al. (2010) | Q | 246 |
| | $1.1\times10^{-1}$ | | Hilal et al. (2008) | Q | |
| | $3.5\times10^{-2}$ | | Modarresi et al. (2007) | Q | 67 |
| | | 4000 | Kühne et al. (2005) | Q | |
| | $1.5\times10^{-1}$ | | Yaffe et al. (2003) | Q | 248, 249 |
| | $1.3\times10^{-1}$ | | English and Carroll (2001) | Q | 230, 231 |
| | $2.4\times10^{-2}$ | | Katritzky et al. (1998) | Q | |
| | $1.3\times10^{-1}$ | | Suzuki et al. (1992) | Q | 232 |
| | $1.4\times10^{-1}$ | | Duchowicz et al. (2020) | ? | 185, 21 |
| | | 3200 | Kühne et al. (2005) | ? | |
| | $1.9\times10^{-1}$ | | Yaws (1999) | ? | 21, 12 |





Table A3.10: Heterocycles with oxygen (. . . continued)

| Substance Formula (Trivial Name) [CAS Registry Number] InChIKey | $H_s^{cp}$ (at $T^\ominus$) $\left[\dfrac{\mathrm{mol}}{\mathrm{m}^3\,\mathrm{Pa}}\right]$ | $\dfrac{\mathrm{d}\ln H_s^{cp}}{\mathrm{d}(1/T)}$ [K] | Reference | Type | Note |
|---|---|---|---|---|---|
| | $1.4\times10^{-1}$ | | Abraham et al. (1990) | ? | |
| tetrahydrofuran-d8 $C_4D_8O$ (THF-d8) [1693-74-9] WYURNTSHIVDZCO-SVYQBANQSA-N | $2.3\times10^{-1}$ | 8000 | Hiatt (2013) | M | |
| 2-methyltetrahydrofuran $CH_3C_4H_7O$ [96-47-9] JWUJQDFVADABEY-UHFFFAOYSA-N | $1.5\times10^{-3}$ | | Mackay et al. (1993) | V | |
| | $1.1\times10^{-1}$ | 6200 | Cabani et al. (1971b) | T | |
| | $1.5\times10^{-1}$ | | Keshavarz et al. (2022) | Q | |
| | $5.3\times10^{-2}$ | | Duchowicz et al. (2020) | Q | |
| | $2.0\times10^{-2}$ | | Raventos-Duran et al. (2010) | Q | 242, 243 |
| | $1.2\times10^{-1}$ | | Raventos-Duran et al. (2010) | Q | 244 |
| | $7.8\times10^{-2}$ | | Raventos-Duran et al. (2010) | Q | 245 |
| | $6.1\times10^{-2}$ | | Hilal et al. (2008) | Q | |
| | $1.7\times10^{-2}$ | | Modarresi et al. (2007) | Q | 67 |
| | | 4400 | Kühne et al. (2005) | Q | |
| | $1.0\times10^{-1}$ | | English and Carroll (2001) | Q | 230, 260 |
| | $2.4\times10^{-2}$ | | Katritzky et al. (1998) | Q | |
| | $9.0\times10^{-2}$ | | Suzuki et al. (1992) | Q | 232 |
| | $1.1\times10^{-1}$ | | Duchowicz et al. (2020) | ? | 185, 21 |
| | | 5400 | Kühne et al. (2005) | ? | |
| 2,5-dimethyltetrahydrofuran $(CH_3)_2C_4H_6O$ [1003-38-9] OXMIDRBAFOEOQT-UHFFFAOYSA-N | $5.5\times10^{-2}$ | 6800 | Cabani et al. (1971b) | T | |
| | $1.6\times10^{-2}$ | | Raventos-Duran et al. (2010) | Q | 271, 243 |
| | $7.8\times10^{-2}$ | | Raventos-Duran et al. (2010) | Q | 244 |
| | $6.2\times10^{-2}$ | | Raventos-Duran et al. (2010) | Q | 245 |
| | $3.1\times10^{-2}$ | | Hilal et al. (2008) | Q | |
| | $2.5\times10^{-2}$ | | Modarresi et al. (2007) | Q | 67 |
| | $7.9\times10^{-2}$ | | English and Carroll (2001) | Q | 230, 231 |
| tetrahydropyran $C_5H_{10}O$ (THP) [142-68-7] DHXVGJBLRPWPCS-UHFFFAOYSA-N | $8.1\times10^{-2}$ | 5800 | Brockbank (2013) | L | 1 |
| | $8.3\times10^{-2}$ | 5900 | Ondo and Dohnal (2007) | M | 1 |
| | $1.0\times10^{-1}$ | | Mackay et al. (2006c) | V | |
| | $1.0\times10^{-1}$ | | Mackay et al. (1993) | V | |
| | $7.8\times10^{-2}$ | 5900 | Cabani et al. (1971b) | T | |
| | $1.5\times10^{-1}$ | | Keshavarz et al. (2022) | Q | |
| | $1.4\times10^{-1}$ | | Duchowicz et al. (2020) | Q | |
| | $2.0\times10^{-2}$ | | Raventos-Duran et al. (2010) | Q | 242, 243 |
| | $2.0\times10^{-1}$ | | Raventos-Duran et al. (2010) | Q | 244 |
| | $7.8\times10^{-2}$ | | Raventos-Duran et al. (2010) | Q | 245 |
| | $1.1\times10^{-1}$ | | Hilal et al. (2008) | Q | |
| | $6.9\times10^{-2}$ | | Modarresi et al. (2007) | Q | 67 |
| | $7.9\times10^{-2}$ | | Yaffe et al. (2003) | Q | 248, 249 |
| | $2.0\times10^{-1}$ | | English and Carroll (2001) | Q | 230, 274 |
| | $2.6\times10^{-2}$ | | Katritzky et al. (1998) | Q | |
| | $9.9\times10^{-2}$ | | Suzuki et al. (1992) | Q | 232 |
| | $7.9\times10^{-2}$ | | Duchowicz et al. (2020) | ? | 185, 21 |





Table A3.10: Heterocycles with oxygen (. . . continued)

| Substance Formula (Trivial Name) [CAS Registry Number] InChIKey | $H_s^{cp}$ (at $T^\ominus$) $\left[\dfrac{\mathrm{mol}}{\mathrm{m^3\,Pa}}\right]$ | $\dfrac{\mathrm{d}\ln H_s^{cp}}{\mathrm{d}(1/T)}$ [K] | Reference | Type | Note |
|---|---|---|---|---|---|
| | $7.9\times10^{-2}$ | | Abraham et al. (1990) | ? | |
| 3-methyltetrahydropyran $C_6H_{12}O$ [26093-63-0] UJQZTMFRMLEYQN-UHFFFAOYSA-N | | 4700 5300 | Kühne et al. (2005) Kühne et al. (2005) | Q ? | |
| 3,4-dihydro-2H-pyran $C_5H_8O$ [110-87-2] BUDQDWGNQVEFAC-UHFFFAOYSA-N | | 3500 3600 | Kühne et al. (2005) Kühne et al. (2005) | Q ? | |
| 1,3-dioxolane $C_3H_6O_2$ [646-06-0] WNXJIVFYUVYPPR-UHFFFAOYSA-N | $4.3\times10^{-1}$ $4.0\times10^{-1}$ $8.1\times10^{-2}$ 2.1 1.5 $2.2\times10^{-1}$ $4.0\times10^{-1}$ | 4800 4800 | Ondo and Dohnal (2007) Cabani et al. (1971b) Keshavarz et al. (2022) Duchowicz et al. (2020) Hilal et al. (2008) Modarresi et al. (2007) Duchowicz et al. (2020) | M T Q Q Q Q ? | 1 184 67 185, 21 |
| 1,3-dioxane $C_4H_8O_2$ [505-22-6] VDFVNEFVBPFDSB-UHFFFAOYSA-N | 1.8 2.1 | | O'Farrell and Waghorne (2010) Hilal et al. (2008) | M Q | |
| 1,4-dioxane $C_4H_8O_2$ (dioxane) [123-91-1] RYHBNJHYFVUHQT-UHFFFAOYSA-N | 1.8 2.3 2.0 2.0 1.4 2.1 4.4 1.4 2.2 1.9 1.1 2.0 1.1 $1.1\times10^{-1}$ 2.8 $7.8\times10^{-1}$ $1.2\times10^1$ 1.6 3.3 1.3 1.5 $8.2\times10^{-1}$ 2.1 | 5800 6600 5800 5100 5800 5200 6100 | Brockbank (2013) Hiatt (2013) Ondo and Dohnal (2007) Welke et al. (1998) Kolb et al. (1992) Park et al. (1987) Ioffe et al. (1984) Friant and Suffet (1979) Rohrschneider (1973) Hwang et al. (1992) Amoore and Buttery (1978) Cabani et al. (1971b) Hayer et al. (2022) Keshavarz et al. (2022) Duchowicz et al. (2020) Raventos-Duran et al. (2010) Raventos-Duran et al. (2010) Raventos-Duran et al. (2010) Hilal et al. (2008) Modarresi et al. (2007) Kühne et al. (2005) English and Carroll (2001) Russell et al. (1992) Duchowicz et al. (2020) Kühne et al. (2005) | L M M M M M M M M V V T Q Q Q Q Q Q Q Q Q Q Q ? ? | 1 1 277 80 38 20 184 242, 243 244 245 67 230, 231 279 185, 21 |



Table A3.10: Heterocycles with oxygen (... continued)

| Substance<br>Formula<br>(Trivial Name)<br>[CAS Registry Number]<br>InChIKey | $H_s^{cp}$<br>(at $T^\ominus$)<br>$\left[\dfrac{\text{mol}}{\text{m}^3\,\text{Pa}}\right]$ | $\dfrac{\text{d}\ln H_s^{cp}}{\text{d}(1/T)}$<br><br>[K] | Reference | Type | Note |
|---|---|---|---|---|---|
| | 1.8 | | Yaws (1999) | ? | 21 |
| | 2.0 | | Betterton (1992) | ? | 533 |
| | 2.2 | | Betterton (1992) | ? | 534 |
| | 1.4 | | Yaws and Yang (1992) | ? | 21 |
| 1,4-dioxane-d8<br>$C_4D_8O_2$<br>(dioxane-d8)<br>[17647-74-4]<br>RYHBNJHYFVUHQT-SVYQBANQSA-N | 2.8 | 6800 | Hiatt (2013) | M | |
| trioxane<br>$C_3H_6O_3$<br>[110-88-3]<br>BGJSXRVXTHVRSN-UHFFFAOYSA-N | $9.6\times10^{-1}$<br>2.7 | | Yaws (2003)<br>Dupeux et al. (2022) | X<br>Q | 258<br>259 |
| 4-methyl-1,3-dioxolan-2-one<br>$C_4H_6O_3$<br>(propylene carbonate)<br>[108-32-7]<br>RUOJZAUFBMNUDX-UHFFFAOYSA-N | $2.9\times10^2$<br>$2.9\times10^2$<br>$1.6\times10^{-1}$<br>$1.4\times10^2$ | | Duchowicz et al. (2020)<br>HSDB (2015)<br>Duchowicz et al. (2020)<br>Abraham et al. (1990) | V<br>V<br>Q<br>? | 186 |
| 1,3,3-trimethyl-2-<br>oxabicyclo[2.2.2]octane<br>$C_{10}H_{18}O$<br>(eucalyptol; 1,8-cineole)<br>[470-82-6]<br>WEEGYLXZBRQIMU-UHFFFAOYSA-N | $5.9\times10^{-2}$<br>$1.2\times10^{-1}$<br>$9.0\times10^{-2}$<br>$7.5\times10^{-2}$<br>$7.4\times10^{-2}$<br>$7.8\times10^{-2}$<br>$5.3\times10^{-3}$<br>$2.2\times10^{-2}$<br>$1.3\times10^{-1}$ | | Kish et al. (2013)<br>Amoore and Buttery (1978)<br>Duchowicz et al. (2020)<br>Copolovici and Niinemets (2005)<br>Niinemets and Reichstein (2002)<br>Amoore and Buttery (1978)<br>Duchowicz et al. (2020)<br>Hilal et al. (2008)<br>Modarresi et al. (2007) | M<br>M<br>V<br>V<br>V<br>V<br>Q<br>Q<br>Q | <br><br>186<br><br><br><br><br><br>67 |
| limonene oxide<br>$C_{10}H_{16}O$<br>[1195-92-2]<br>CCEFMUBVSUDRLG-UHFFFAOYSA-N | $5.6\times10^{-2}$<br>$5.5\times10^{-2}$<br>$2.7\times10^{-2}$<br>$4.8\times10^{-2}$ | <br><br>4600 | Fichan et al. (1999)<br>Duchowicz et al. (2020)<br>van Roon et al. (2005)<br>Duchowicz et al. (2020) | M<br>V<br>V<br>Q | <br>186 |
| dibenzo[$b,e$][1,4]dioxin<br>$C_{12}H_8O_2$<br>(dibenzo-$p$-dioxin)<br>[262-12-4]<br>NFBOHOGPQUYFRF-UHFFFAOYSA-N | $8.9\times10^{-2}$<br>$9.0\times10^{-2}$<br>$8.5\times10^{-2}$<br>$9.5\times10^{-3}$<br>$8.5\times10^{-2}$<br>$8.1\times10^{-2}$<br>2.0<br>$2.7\times10^{-2}$<br>$6.3\times10^{-2}$<br>$9.1\times10^{-2}$ | | Duchowicz et al. (2020)<br>HSDB (2015)<br>Mackay et al. (2006b)<br>Saçan et al. (2005)<br>Govers and Krop (1998)<br>Shiu et al. (1988)<br>Duchowicz et al. (2020)<br>Saçan et al. (2005)<br>Wang and Wong (2002)<br>Govers and Krop (1998) | V<br>V<br>V<br>V<br>V<br>V<br>Q<br>Q<br>Q<br>Q | 186<br><br><br><br><br><br><br><br>535 |



Table A3.10: Heterocycles with oxygen (...continued)

| Substance<br>Formula<br>(Trivial Name)<br>[CAS Registry Number]<br>InChIKey | $H_s^{cp}$<br>(at $T^\ominus$)<br>$\left[\dfrac{\mathrm{mol}}{\mathrm{m^3\,Pa}}\right]$ | $\dfrac{\mathrm{d\ln}H_s^{cp}}{\mathrm{d}(1/T)}$<br><br>[K] | Reference | Type | Note |
|---|---|---|---|---|---|
| piperonal<br>$C_8H_6O_3$<br>[120-57-0]<br>SATCULPHIDQDRE-UHFFFAOYSA-N | $1.8\times10^1$<br>$1.8\times10^1$<br>$1.6\times10^2$<br>$4.1\times10^2$<br>$1.1\times10^1$ | | Duchowicz et al. (2020)<br>HSDB (2015)<br>Duchowicz et al. (2020)<br>Hilal et al. (2008)<br>Modarresi et al. (2007) | V<br>V<br>Q<br>Q<br>Q | 186<br><br><br><br>67 |
| paraldehyde<br>$C_6H_{12}O_3$<br>[123-63-7]<br>SQYNKIJPMDEDEG-UHFFFAOYSA-N | $5.8\times10^{-1}$<br>$2.5\times10^{-1}$<br>4.7<br>$3.6\times10^{-1}$<br>2.1<br>$6.4\times10^{-1}$ | | Duchowicz et al. (2020)<br>HSDB (2015)<br>Duchowicz et al. (2020)<br>Hilal et al. (2008)<br>Modarresi et al. (2007)<br>Yaws (1999) | V<br>V<br>Q<br>Q<br>Q<br>? | 186<br><br><br><br>67<br>21, 38 |
| benzofuran<br>$C_8H_6O$<br>[271-89-6]<br>IANQTJSKSUMEQM-UHFFFAOYSA-N | $1.9\times10^{-2}$<br>$1.9\times10^{-2}$ | | HSDB (2015)<br>Hilal et al. (2008) | Q<br>Q | 99 |
| $\gamma$-nonalactone<br>$C_9H_{16}O_2$<br>[104-61-0]<br>OALYTRUKMRCXNH-UHFFFAOYSA-N | $1.8\times10^{-1}$ | | Hertel and Sommer (2006) | Q | 415 |
| xanthene<br>$C_{13}H_{10}O$<br>[92-83-1]<br>GJCOSYZMQJWQCA-UHFFFAOYSA-N | $1.3\times10^{-1}$ | | Abraham et al. (2019) | Q | |
| 1,5,5,9-tetramethyl-13-<br>oxatricyclo(8.3.0.0(4,9))tridecane<br>$C_{16}H_{28}O$<br>(ambroxan)<br>[3738-00-9]<br>YPZUZOLGGMJZJO-UHFFFAOYSA-N | $2.0\times10^{-2}$<br><br>$2.9\times10^{-1}$<br>$6.5\times10^{-2}$<br>$1.1\times10^{-3}$ | | Zhang et al. (2010)<br><br>Zhang et al. (2010)<br>Zhang et al. (2010)<br>Zhang et al. (2010) | Q<br><br>Q<br>Q<br>Q | 287, 288<br><br>287, 289<br>287, 290<br>287, 291 |
| 1,3,4,6,7,8-hexahydro-4,6,6,7,8,8-<br>hexamethylcyclopenta[$g$]-2-<br>benzopyran<br>$C_{18}H_{26}O$<br>[1222-05-5]<br>ONKNPOPIGWHAQC-UHFFFAOYSA-N | $7.6\times10^{-2}$<br><br><br>$7.5\times10^{-2}$<br>8.2<br>$8.4\times10^{-2}$<br>$9.9\times10^{-3}$ | | HSDB (2015)<br><br><br>Zhang et al. (2010)<br>Zhang et al. (2010)<br>Zhang et al. (2010)<br>Zhang et al. (2010) | V<br><br><br>Q<br>Q<br>Q<br>Q | <br><br><br>287, 288<br>287, 289<br>287, 290<br>287, 291 |
| cinmethylin<br>$C_{18}H_{26}O_2$<br>[87818-31-3]<br>QMTNOLKHSWIQBE-FGTMMUONSA-N | $1.3\times10^1$ | | Ebert et al. (2023) | ? | 318 |



Table A3.10: Heterocycles with oxygen (... continued)

| Substance<br>Formula<br>(Trivial Name)<br>[CAS Registry Number]<br>InChIKey | $H_s^{cp}$<br>(at $T^{\ominus}$)<br><br>$\left[\dfrac{\text{mol}}{\text{m}^3\,\text{Pa}}\right]$ | $\dfrac{\text{d}\ln H_s^{cp}}{\text{d}(1/T)}$<br><br>[K] | Reference | Type | Note |
|---|---|---|---|---|---|
| milbemycin A4<br>$C_{32}H_{46}O_7$<br>(milbemectin A4)<br>[51596-11-3]<br>VOZIAWLUULBIPN-LRBNAKOISA-N | $7.7\times10^6$ | | Ebert et al. (2023) | ? | 318 |



### A3.11  Oxidized terpenoids

Table A3.11: Oxidized terpenoids

| Substance Formula (Trivial Name) [CAS Registry Number] InChIKey | $H_s^{cp}$ (at $T^{\ominus}$) $\left[\dfrac{\text{mol}}{\text{m}^3\,\text{Pa}}\right]$ | $\dfrac{\text{d}\ln H_s^{cp}}{\text{d}(1/T)}$ [K] | Reference | Type | Note |
|---|---|---|---|---|---|
| (1S-endo)-1,7,7-trimethyl-bicyclo[2.2.1]heptan-2-ol | $4.5\times10^{-1}$ | | Fichan et al. (1999) | M | |
| $C_{10}H_{18}O$ | $7.2\times10^{-1}$ | | Duchowicz et al. (2020) | V | 186 |
| (1S-endo-(-)-borneol) | $4.0\times10^{-1}$ | | Duchowicz et al. (2020) | Q | |
| [464-45-9] | | | | | |
| DTGKSKDOIYIVQL-QXFUBDJGSA-N | | | | | |
| (1R)-1,3,3-trimethylbicyclo[2.2.1]heptan-2-ol | $3.6\times10^{-1}$ | | Fichan et al. (1999) | M | |
| $C_{10}H_{18}O$ | $3.6\times10^{-1}$ | | Duchowicz et al. (2020) | V | 186 |
| (endo-(+)-fenchyl alcohol) | $4.0\times10^{-1}$ | | Duchowicz et al. (2020) | Q | |
| [2217-02-9] | | | | | |
| IAIHUHQCLTYTSF-OYNCUSHFSA-N | | | | | |
| 2-(4-methyl-3-cyclohexen-1-yl)-2-propanol | 4.4 | 2200 | Copolovici and Niinemets (2005) | M | |
| $C_{10}H_{18}O$ | 4.1 | | Copolovici and Niinemets (2005) | V | |
| ($\alpha$-terpineol) | $6.0\times10^{-1}$ | 4800 | van Roon et al. (2005) | V | |
| [98-55-5] | 4.2 | | Niinemets and Reichstein (2002) | V | |
| WUOACPNHFRMFPN-UHFFFAOYSA-N | $7.4\times10^{-1}$ | 5400 | Li et al. (1998) | V | |
| | 6.5 | | Dupeux et al. (2022) | Q | 259 |
| | 3.6 | | Hilal et al. (2008) | Q | |
| | $8.2\times10^{-1}$ | | Modarresi et al. (2007) | Q | 67 |
| 1,2-dimethyl-3-(1-methylethenyl)-cyclopentanol | $5.3\times10^{-1}$ | | Duchowicz et al. (2020) | V | 186 |
| $C_{10}H_{18}O$ | $4.0\times10^{-1}$ | 17000 | Li et al. (1998) | V | |
| (plinol) | $3.4\times10^{-1}$ | | Duchowicz et al. (2020) | Q | |
| [72402-00-7] | 1.2 | | Hilal et al. (2008) | Q | |
| ZRVPDCMGGOSDKG-UHFFFAOYSA-N | 1.6 | | Modarresi et al. (2007) | Q | 67 |
| 1-methyl-4-(1-methylethyl)-7-oxabicyclo[2.2.1]heptane | $3.9\times10^{-2}$ | | Helburn et al. (2008) | M | |
| $C_{10}H_{18}O$ | $7.4\times10^{-2}$ | | Copolovici and Niinemets (2005) | V | |
| (1,4-cineole) | $1.4\times10^{-1}$ | 4000 | van Roon et al. (2005) | V | |
| [470-67-7] | | | | | |
| RFFOTVCVTJUTAD-UHFFFAOYSA-N | | | | | |
| 1,7,7-trimethyl-bicyclo[2.2.1]heptan-2-one | $1.2\times10^{-1}$ | | Duchowicz et al. (2020) | V | 186 |
| $C_{10}H_{16}O$ | $1.2\times10^{-1}$ | | HSDB (2015) | V | |
| (camphor) | 1.1 | | Copolovici and Niinemets (2005) | V | |
| [76-22-2] | $5.4\times10^{-1}$ | 4800 | van Roon et al. (2005) | V | |
| DSSYKIVIOFKYAU-UHFFFAOYSA-N | $8.2\times10^{-1}$ | | Niinemets and Reichstein (2002) | V | |
| | $2.3\times10^{-2}$ | | Duchowicz et al. (2020) | Q | |
| | $3.5\times10^{-1}$ | | Modarresi et al. (2007) | Q | 67 |



Table A3.11: Oxidized terpenoids (... continued)

| Substance Formula (Trivial Name) [CAS Registry Number] InChIKey | $H_s^{cp}$ (at $T^{\ominus}$) $\left[\dfrac{\text{mol}}{\text{m}^3\,\text{Pa}}\right]$ | $\dfrac{\text{d}\ln H_s^{cp}}{\text{d}(1/T)}$ [K] | Reference | Type | Note |
|---|---|---|---|---|---|
| 2,7,7-trimethyl-3-oxatricyclo[4.1.1.0(2,4)]octane | $2.3\times10^{-2}$ | | Fichan et al. (1999) | M | |
| $C_{10}H_{16}O$ | $2.3\times10^{-2}$ | | Duchowicz et al. (2020) | V | 186 |
| ((-)-$\alpha$-pinene oxide) | $2.4\times10^{-2}$ | | Copolovici and Niinemets (2005) | V | |
| [1686-14-2] | $5.4\times10^{-2}$ | 4400 | van Roon et al. (2005) | V | |
| NQFUSWIGRKFAHK-UHFFFAOYSA-N | $2.2\times10^{-2}$ | | Duchowicz et al. (2020) | Q | |
| 5-methyl-2-(1-methylethylidene)-cyclohexanone | $2.8\times10^{-1}$ | 5300 | van Roon et al. (2005) | V | |
| $C_{10}H_{16}O$ | $1.7\times10^{-1}$ | | HSDB (2015) | Q | 99 |
| (pulegone) | | | | | |
| [89-82-7] | | | | | |
| NZGWDASTMWDZIW-UHFFFAOYSA-N | | | | | |
| exo-2-[(1,7,7-trimethylbicyclo[2.2.1]hept-2-yl)-oxy]ethanol | 1.0 | 4100 | Li et al. (1998) | V | |
| $C_{12}H_{22}O_2$ | | | | | |
| (arbanol) | | | | | |
| [7070-15-7] | | | | | |
| IWWCSDGEIDYEJV-JBLDHEPKSA-N | | | | | |
| bornyl acetate | $3.8\times10^{-4}$ | 1700 | Copolovici and Niinemets (2015) | M | |
| $C_{12}H_{20}O_2$ | | | | | |
| [5655-61-8] | | | | | |
| KGEKLUUHTZCSIP-SCVCMEIPSA-N | | | | | |
| $\beta$-ionone | 1.2 | | Fichan et al. (1999) | M | |
| $C_{13}H_{20}O$ | $1.2\times10^{-1}$ | | Duchowicz et al. (2020) | V | 186 |
| [79-77-6] | $1.5\times10^{-2}$ | | Abney (2021) | Q | 399 |
| PSQYTAPXSHCGMF-BQYQJAHWSA-N | $6.6\times10^{-3}$ | | Duchowicz et al. (2020) | Q | |
| nerolidol | $3.2\times10^{-4}$ | 4300 | Copolovici and Niinemets (2015) | M | |
| $C_{15}H_{26}O$ | | | | | |
| [7212-44-4] | | | | | |
| FQTLCLSUCSAZDY-UHFFFAOYSA-N | | | | | |



## A3.12   Miscellaneous

Table A3.12: Miscellaneous

| Substance Formula (Trivial Name) [CAS Registry Number] InChIKey | $H_s^{cp}$ (at $T^{\ominus}$) $\left[\dfrac{\text{mol}}{\text{m}^3\,\text{Pa}}\right]$ | $\dfrac{\text{d}\ln H_s^{cp}}{\text{d}(1/T)}$ [K] | Reference | Type | Note |
|---|---|---|---|---|---|
| oxoethanoic acid | $1.1\times10^2$ | 4800 | Burkholder et al. (2019) | L | 460 |
| OHCCOOH | $1.1\times10^2$ | 4800 | Burkholder et al. (2015) | L | 460 |
| (glyoxylic acid) | $1.1\times10^2$ | 4800 | Sander et al. (2011) | L | |
| [298-12-4] | $1.1\times10^2$ | 4800 | Ip et al. (2009) | M | |
| HHLFWLYXYJOTON-UHFFFAOYSA-N | $1.3\times10^4$ | | Wang et al. (2017) | Q | 80, 238 |
| | $1.5\times10^1$ | | Wang et al. (2017) | Q | 80, 239 |
| | $6.9\times10^1$ | | Wang et al. (2017) | Q | 80, 240 |
| | $3.3\times10^3$ | | HSDB (2015) | Q | 99 |
| | $7.8\times10^4$ | | Raventos-Duran et al. (2010) | Q | 242, 243 |
| | $2.5\times10^8$ | | Raventos-Duran et al. (2010) | Q | 244 |
| | $3.1\times10^3$ | | Raventos-Duran et al. (2010) | Q | 245 |
| | $8.9\times10^1$ | | Saxena and Hildemann (1996) | E | 401 |
| | | | Warneck (2005) | ? | 536 |
| hydroxyethanoic acid | $2.8\times10^2$ | 4000 | Burkholder et al. (2019) | L | 460, 537 |
| HOCH$_2$COOH | $2.8\times10^2$ | 4000 | Burkholder et al. (2015) | L | 460, 538 |
| (glycolic acid) | $2.8\times10^2$ | 4000 | Sander et al. (2011) | L | |
| [79-14-1] | $2.8\times10^2$ | 4000 | Ip et al. (2009) | M | |
| AEMRFAOFKBGASW-UHFFFAOYSA-N | $1.9\times10^3$ | | Wang et al. (2017) | Q | 80, 238 |
| | $1.4\times10^5$ | | Wang et al. (2017) | Q | 80, 239 |
| | $5.3\times10^4$ | | Wang et al. (2017) | Q | 80, 240 |
| | $1.2\times10^3$ | | Raventos-Duran et al. (2010) | Q | 242, 243 |
| | $3.1\times10^4$ | | Raventos-Duran et al. (2010) | Q | 244 |
| | $1.2\times10^2$ | | Raventos-Duran et al. (2010) | Q | 245 |
| 2-hydroxyethanal | $3.9\times10^2$ | 4600 | Burkholder et al. (2019) | L | 460 |
| HOCH$_2$CHO | $3.9\times10^2$ | 4600 | Burkholder et al. (2015) | L | 460 |
| (hydroxyacetaldehyde; glycolaldehyde) | $4.1\times10^2$ | 4600 | Betterton and Hoffmann (1988) | M | 460 |
| [141-46-8] | $9.9\times10^2$ | | Lee and Zhou (1993) | C | 87 |
| WGCNASOHLSPBMP-UHFFFAOYSA-N | $2.9\times10^2$ | | Keshavarz et al. (2022) | Q | |
| | $1.8\times10^1$ | | Duchowicz et al. (2020) | Q | 299 |
| | $1.3\times10^1$ | | Wang et al. (2017) | Q | 80, 238 |
| | $1.1\times10^2$ | | Wang et al. (2017) | Q | 80, 239 |
| | 4.2 | | Wang et al. (2017) | Q | 80, 240 |
| | $3.1\times10^1$ | | Raventos-Duran et al. (2010) | Q | 271, 243 |
| | $7.8\times10^3$ | | Raventos-Duran et al. (2010) | Q | 244 |
| | $9.9\times10^{-1}$ | | Raventos-Duran et al. (2010) | Q | 245 |
| | $6.5\times10^2$ | | Hilal et al. (2008) | Q | |
| | $2.4\times10^2$ | | Modarresi et al. (2007) | Q | 67 |
| | | 7600 | Kühne et al. (2005) | Q | |
| | $4.1\times10^2$ | | Duchowicz et al. (2020) | ? | 185, 21 |
| | | 4600 | Kühne et al. (2005) | ? | |



Table A3.12: Miscellaneous (...continued)

| Substance Formula (Trivial Name) [CAS Registry Number] InChIKey | $H_s^{cp}$ (at $T^\ominus$) $\left[\dfrac{\mathrm{mol}}{\mathrm{m^3\,Pa}}\right]$ | $\dfrac{\mathrm{d}\ln H_s^{cp}}{\mathrm{d}(1/T)}$ [K] | Reference | Type | Note |
|---|---|---|---|---|---|
| propanonal | $3.5\times10^1$ | 7500 | Burkholder et al. (2019) | L | 460 |
| $CH_3COCHO$ | $3.5\times10^1$ | 7500 | Burkholder et al. (2015) | L | 460 |
| (methylglyoxal; pyruvaldehyde) | $3.2\times10^2$ | | Zhou and Mopper (1990) | M | 70 |
| [78-98-8] | $3.4\times10^1$ | 7500 | Betterton and Hoffmann (1988) | M | 460 |
| AIJULSRZWUXGPQ-UHFFFAOYSA-N | $3.7\times10^2$ | | Lee and Zhou (1993) | C | 87 |
| | $5.4\times10^1$ | | Wang et al. (2017) | Q | 80, 238 |
| | $1.3\times10^1$ | | Wang et al. (2017) | Q | 80, 239 |
| | $2.9\times10^{-2}$ | | Wang et al. (2017) | Q | 80, 240 |
| | $6.2\times10^2$ | | Raventos-Duran et al. (2010) | Q | 271, 243 |
| | $4.9\times10^2$ | | Raventos-Duran et al. (2010) | Q | 244 |
| | $3.9\times10^1$ | | Raventos-Duran et al. (2010) | Q | 245 |
| | $1.4\times10^1$ | | Modarresi et al. (2007) | Q | 67 |
| | | 6200 | Kühne et al. (2005) | Q | |
| | 8.2 | | Katritzky et al. (1998) | Q | |
| | | 7600 | Kühne et al. (2005) | ? | |
| 2-oxopropanoic acid | $3.1\times10^3$ | 5100 | Burkholder et al. (2019) | L | |
| $CH_3COCOOH$ | $3.1\times10^3$ | 5100 | Burkholder et al. (2015) | L | |
| (pyruvic acid) | $3.1\times10^3$ | 5100 | Sander et al. (2011) | L | |
| [127-17-3] | $3.1\times10^3$ | 5100 | Sander et al. (2006) | L | |
| LCTONWCANYUPML-UHFFFAOYSA-N | $3.0\times10^3$ | 5300 | Staudinger and Roberts (2001) | L | |
| | $3.1\times10^3$ | 5100 | Khan et al. (1995) | M | |
| | $3.1\times10^3$ | | Khan et al. (1992) | M | |
| | $3.1\times10^3$ | 5200 | Khan and Brimblecombe (1992) | M | |
| | $2.2\times10^4$ | | Keshavarz et al. (2022) | Q | |
| | $2.3\times10^2$ | | Duchowicz et al. (2020) | Q | 184 |
| | $7.8\times10^3$ | | Wang et al. (2017) | Q | 80, 238 |
| | $5.0\times10^1$ | | Wang et al. (2017) | Q | 80, 239 |
| | $1.9\times10^1$ | | Wang et al. (2017) | Q | 80, 240 |
| | $1.6\times10^3$ | | Raventos-Duran et al. (2010) | Q | 242, 243 |
| | $2.0\times10^6$ | | Raventos-Duran et al. (2010) | Q | 244 |
| | $4.9\times10^3$ | | Raventos-Duran et al. (2010) | Q | 245 |
| | $2.6\times10^3$ | | Hilal et al. (2008) | Q | |
| | $2.6\times10^3$ | | Modarresi et al. (2007) | Q | 67 |
| | | 5600 | Kühne et al. (2005) | Q | |
| | $3.1\times10^3$ | | Duchowicz et al. (2020) | ? | 185, 21 |
| | | 5300 | Kühne et al. (2005) | ? | |
| 3-oxopropanoic acid | $1.0\times10^4$ | | Wang et al. (2017) | Q | 80, 238 |
| $OHCCH_2COOH$ | $1.2\times10^4$ | | Wang et al. (2017) | Q | 80, 239 |
| [926-61-4] | $1.7\times10^4$ | | Wang et al. (2017) | Q | 80, 240 |
| OAKURXIZZOAYBC-UHFFFAOYSA-N | $6.9\times10^1$ | | Saxena and Hildemann (1996) | E | 401 |



Table A3.12: Miscellaneous (...continued)

| Substance Formula (Trivial Name) [CAS Registry Number] InChIKey | $H_s^{cp}$ (at $T^{\ominus}$) $\left[\dfrac{\text{mol}}{\text{m}^3\,\text{Pa}}\right]$ | $\dfrac{\text{d}\ln H_s^{cp}}{\text{d}(1/T)}$ [K] | Reference | Type | Note |
|---|---|---|---|---|---|
| 2-hydroxypropanoic acid | $1.2\times10^2$ | | Duchowicz et al. (2020) | V | 186 |
| CH$_3$CHOHCOOH | $1.2\times10^2$ | | HSDB (2015) | V | |
| (lactic acid) | $1.1\times10^3$ | | Duchowicz et al. (2020) | Q | |
| [50-21-5] | $9.9\times10^2$ | | Raventos-Duran et al. (2010) | Q | 242, 243 |
| JVTAAEKCZFNVCJ-UHFFFAOYSA-N | $9.9\times10^3$ | | Raventos-Duran et al. (2010) | Q | 244 |
| | $9.9\times10^1$ | | Raventos-Duran et al. (2010) | Q | 245 |
| | $6.9\times10^5$ | | Saxena and Hildemann (1996) | E | 401 |
| glycidaldehyde | $1.9\times10^1$ | | HSDB (2015) | Q | 99 |
| C$_3$H$_4$O$_2$ | | | | | |
| [765-34-4] | | | | | |
| IWYRWIUNAVNFPE-UHFFFAOYSA-N | | | | | |
| trimethylene oxide | $4.0\times10^{-1}$ | | Duchowicz et al. (2020) | V | 186 |
| C$_3$H$_6$O | $3.9\times10^{-1}$ | | HSDB (2015) | V | |
| (1,3-epoxypropane; 1,3-propylene oxide; oxetane) | $1.2\times10^{-1}$ | | Duchowicz et al. (2020) | Q | |
| [503-30-0] | | | | | |
| AHHWIHXENZJRFG-UHFFFAOYSA-N | | | | | |
| 2,3-dihydroxypropanal | $1.4\times10^4$ | | Wang et al. (2017) | Q | 80, 238 |
| C$_3$H$_6$O$_3$ | $2.6\times10^4$ | | Wang et al. (2017) | Q | 80, 239 |
| (glyceraldehyde) | $8.1\times10^2$ | | Wang et al. (2017) | Q | 80, 240 |
| [367-47-5] | $2.0\times10^8$ | | Saxena and Hildemann (1996) | E | 401 |
| MNQZXJOMYWMBOU-UHFFFAOYSA-N | | | | | |
| dihydroxyacetone | $1.8\times10^6$ | | HSDB (2015) | V | |
| C$_3$H$_6$O$_3$ | $5.5\times10^3$ | | Wang et al. (2017) | Q | 80, 238 |
| [96-26-4] | $3.8\times10^3$ | | Wang et al. (2017) | Q | 80, 239 |
| RXKJFZQQPQGTFL-UHFFFAOYSA-N | $5.0\times10^3$ | | Wang et al. (2017) | Q | 80, 240 |
| 2-methoxyethanol | 4.4 | 7500 | Hiatt (2013) | M | |
| C$_3$H$_8$O$_2$ | $2.2\times10^{-4}$ | -730 | Ashworth et al. (1988) | M | 539, 42, 278 |
| (methyl cellosolve) | $1.4\times10^1$ | | Johanson and Dynésius (1988) | M | 14 |
| [109-86-4] | $3.7\times10^1$ | 7300 | Cabani et al. (1978) | T | |
| XNWFRZJHXBZDAG-UHFFFAOYSA-N | 6.8 | | Keshavarz et al. (2022) | Q | |
| | $1.4\times10^1$ | | Duchowicz et al. (2020) | Q | 184 |
| | 8.9 | | Wang et al. (2017) | Q | 80, 238 |
| | $5.4\times10^1$ | | Wang et al. (2017) | Q | 80, 239 |
| | $1.3\times10^1$ | | Wang et al. (2017) | Q | 80, 240 |
| | $3.1\times10^1$ | | Raventos-Duran et al. (2010) | Q | 242, 243 |
| | $2.0\times10^1$ | | Raventos-Duran et al. (2010) | Q | 244 |
| | $2.5\times10^2$ | | Raventos-Duran et al. (2010) | Q | 245 |
| | $2.1\times10^1$ | | Hilal et al. (2008) | Q | |
| | $4.0\times10^1$ | | Modarresi et al. (2007) | Q | 67 |
| | $1.5\times10^1$ | | Nirmalakhandan et al. (1997) | Q | |
| | $3.0\times10^1$ | | Duchowicz et al. (2020) | ? | 185, 21 |



Table A3.12: Miscellaneous (...continued)

| Substance Formula (Trivial Name) [CAS Registry Number] InChIKey | $H_s^{cp}$ (at $T^{\ominus}$) $\left[\dfrac{\text{mol}}{\text{m}^3\,\text{Pa}}\right]$ | $\dfrac{\text{d}\ln H_s^{cp}}{\text{d}(1/T)}$ [K] | Reference | Type | Note |
|---|---|---|---|---|---|
| 4-oxobutanoic acid | $8.3\times10^3$ | | Wang et al. (2017) | Q | 80, 238 |
| $OHC(CH_2)_2COOH$ | $3.2\times10^5$ | | Wang et al. (2017) | Q | 80, 239 |
| [692-29-5] | $2.1\times10^4$ | | Wang et al. (2017) | Q | 80, 240 |
| UIUJIQZEACWQSV-UHFFFAOYSA-N | $4.9\times10^1$ | | Saxena and Hildemann (1996) | E | 401 |
| 2,3-dihydroxybutanedioic acid | $4.9\times10^{15}$ | | Burkholder et al. (2019) | L | |
| HOOCCHOHCHOHCOOH | $4.9\times10^{15}$ | | Burkholder et al. (2015) | L | |
| (tartaric acid) | | | Compernolle and Müller (2014a) | V | 540 |
| [87-69-4] | $1.3\times10^{13}$ | | Yaws (2003) | X | 237, 12 |
| FEWJPZIEWOKRBE-JCYAYHJZSA-N | $5.1\times10^{10}$ | | Gharagheizi et al. (2012) | Q | |
| | $1.3\times10^{13}$ | | Gharagheizi et al. (2010) | Q | 246 |
| | $9.9\times10^{15}$ | | Saxena and Hildemann (1996) | E | 401 |
| 3-oxapentane-1,5-diol | $3.3\times10^4$ | | Yaws (2003) | X | 258 |
| $HO(CH_2)_2O(CH_2)_2OH$ | $3.3\times10^4$ | | Yaws (2003) | X | 237 |
| (diethylene glycol) | $9.0\times10^3$ | | Dupeux et al. (2022) | Q | 259 |
| [111-46-6] | $2.5\times10^3$ | | Olsen et al. (2016) | Q | 425 |
| MTHSVFCYNBDYFN-UHFFFAOYSA-N | $1.6\times10^3$ | | Olsen et al. (2016) | Q | 426 |
| | $1.7\times10^3$ | | Olsen et al. (2016) | Q | 427 |
| | $4.9\times10^3$ | | HSDB (2015) | Q | 99 |
| | $7.7\times10^3$ | | Gharagheizi et al. (2012) | Q | |
| | $3.8\times10^4$ | | Gharagheizi et al. (2010) | Q | 246 |
| | $2.4\times10^4$ | | Hilal et al. (2008) | Q | |
| | $2.2\times10^4$ | | Yao et al. (2002) | Q | 229 |
| | $2.0\times10^7$ | | Saxena and Hildemann (1996) | E | 401 |
| | $3.3\times10^4$ | | Yaws (1999) | ? | 21 |
| hydroxybutanedioic acid | $2.7\times10^8$ | | Compernolle and Müller (2014a) | V | |
| $HOOCCH_2CHOHCOOH$ | $1.7\times10^9$ | | Yaws (2003) | X | 237, 372 |
| (malic acid) | $1.2\times10^7$ | | HSDB (2015) | Q | 99 |
| [6915-15-7] | $5.3\times10^8$ | | Gharagheizi et al. (2012) | Q | |
| BJEPYKJPYRNKOW-UHFFFAOYSA-N | $1.4\times10^9$ | | Gharagheizi et al. (2010) | Q | 246 |
| | $2.0\times10^{11}$ | | Saxena and Hildemann (1996) | E | 401 |
| 2-ethoxyethanol | $8.9$ | | Johanson and Dynésius (1988) | M | 14 |
| $C_4H_{10}O_2$ | $3.3\times10^1$ | | Abraham et al. (1994a) | R | |
| [110-80-5] | $2.8\times10^1$ | 8000 | Cabani et al. (1978) | T | |
| ZNQVEEAIQZEUHB-UHFFFAOYSA-N | $9.2$ | | Keshavarz et al. (2022) | Q | |
| | $5.8$ | | Duchowicz et al. (2020) | Q | 299 |
| | $7.4$ | | Wang et al. (2017) | Q | 80, 238 |
| | $3.4\times10^1$ | | Wang et al. (2017) | Q | 80, 239 |
| | $9.8$ | | Wang et al. (2017) | Q | 80, 240 |
| | $2.5\times10^1$ | | Raventos-Duran et al. (2010) | Q | 242, 243 |
| | $1.2\times10^1$ | | Raventos-Duran et al. (2010) | Q | 244 |
| | $1.6\times10^2$ | | Raventos-Duran et al. (2010) | Q | 245 |
| | $1.6\times10^1$ | | Hilal et al. (2008) | Q | |
| | $3.2\times10^1$ | | Modarresi et al. (2007) | Q | 67 |
| | $7.5$ | | Nirmalakhandan et al. (1997) | Q | |
| | $2.1\times10^1$ | | Duchowicz et al. (2020) | ? | 185, 21 |



Table A3.12: Miscellaneous (. . . continued)

| Substance Formula (Trivial Name) [CAS Registry Number] InChIKey | $H_s^{cp}$ (at $T^\ominus$) $\left[\dfrac{\text{mol}}{\text{m}^3\,\text{Pa}}\right]$ | $\dfrac{\mathrm{d}\ln H_s^{cp}}{\mathrm{d}(1/T)}$ [K] | Reference | Type | Note |
|---|---|---|---|---|---|
| | $1.9\times10^1$ | | Yaws (1999) | ? | 21, 12 |
| 2-methoxy-1-propanol $C_4H_{10}O_2$ [1589-47-5] YTTFFPATQICAQN-UHFFFAOYSA-N | $5.5\times10^2$ | | HSDB (2015) | Q | 99 |
| 1,1-dimethoxyethane $C_4H_{10}O_2$ [534-15-6] SPEUIVXLLWOEMJ-UHFFFAOYSA-N | $1.5\times10^{-1}$ | | HSDB (2015) | Q | 99 |
| 4-methylene-2-oxetanone $C_4H_4O_2$ (acetyl ketene) [674-82-8] WASQWSOJHCZDFK-UHFFFAOYSA-N | $1.6\times10^{-2}$ | | HSDB (2015) | Q | 99 |
| 2(5H)-furanone $C_4H_4O_2$ [497-23-4] VIHAEDVKXSOUAT-UHFFFAOYSA-N | $3.5\times10^{-1}$ $3.6\times10^2$ $1.4\times10^3$ $1.0$ | | Wang et al. (2017) Wang et al. (2017) Wang et al. (2017) HSDB (2015) | Q Q Q Q | 80, 238 80, 239 80, 240 99 |
| 2,2'-bioxirane $C_4H_6O_2$ [1464-53-5] ZFIVKAOQEXOYFY-UHFFFAOYSA-N | $2.8\times10^2$ | | HSDB (2015) | Q | 99 |
| $\gamma$-butyrolactone $C_4H_6O_2$ [96-48-0] YEJRWHAVMIAJKC-UHFFFAOYSA-N | $1.9\times10^2$ $1.9\times10^2$ $6.5\times10^1$ $1.2$ | | Duchowicz et al. (2020) HSDB (2015) Dupeux et al. (2022) Duchowicz et al. (2020) | V V Q Q | 186  259  |
| ethyloxirane $C_4H_8O$ (1,2-epoxybutane) [106-88-7] RBACIKXCRWGCBB-UHFFFAOYSA-N | $5.5\times10^{-2}$ $5.5\times10^{-2}$ $6.9\times10^{-2}$ | | Duchowicz et al. (2020) HSDB (2015) Duchowicz et al. (2020) | V V Q | 186  |
| 2,3-epoxy-2-methyl-1,4-butanediol $C_5H_{10}O_3$ (IEPOX) FLVAIUDBQNOKHB-UHFFFAOYSA-N | $5.0\times10^4$ $1.3\times10^5$ $3.2\times10^4$ $2.7\times10^4$  $1.3\times10^6$ $1.9\times10^5$ $3.0\times10^5$ | | Wang et al. (2017) Wang et al. (2017) Wang et al. (2017) Pye et al. (2013) Chan et al. (2010) Eddingsaas et al. (2010) Vasilakos et al. (2021) Woo and McNeill (2015) | Q Q Q Q Q Q E ? | 80, 238 80, 239 80, 240 492 541 542 543 466 |
| MEDICO1CO $C_5H_8O_4$ RRXXDUUNVWXRFN-UHFFFAOYSA-N | $2.5\times10^7$ | 12000 | Wieser et al. (2023) | Q | 437 |



Table A3.12: Miscellaneous (...continued)

| Substance / Formula / (Trivial Name) / [CAS Registry Number] / InChIKey | $H_s^{cp}$ (at $T^{\ominus}$) $\left[\dfrac{\text{mol}}{\text{m}^3\,\text{Pa}}\right]$ | $\dfrac{\text{d}\ln H_s^{cp}}{\text{d}(1/T)}$ [K] | Reference | Type | Note |
|---|---|---|---|---|---|
| MEDICO4CO<br>$C_5H_8O_4$<br>GXKSWJOBVXKANI-UHFFFAOYSA-N | $3.1\times10^6$ | 12000 | Wieser et al. (2023) | Q | 437 |
| METRICO<br>$C_5H_{10}O_4$<br>LCGBCDAYKOJPSO-UHFFFAOYSA-N | $7.3\times10^6$ | 12000 | Wieser et al. (2023) | Q | 437 |
| 2,3-epoxy-6-oxo-heptenal<br>$C_7H_8O_3$<br>(MCM:TLEPOXMUC)<br>YHZQHUOBDYTWMQ-UHFFFAOYSA-N | $1.1\times10^4$<br>$1.4\times10^4$<br>$2.8\times10^3$<br>$2.5\times10^3$ | | Wang et al. (2017)<br>Wang et al. (2017)<br>Wang et al. (2017)<br>McNeill et al. (2012) | Q<br>Q<br>Q<br>Q | 80, 238<br>80, 239<br>80, 240 |
| 3-hydroxy-2-butanone<br>$C_4H_8O_2$<br>(acetoin)<br>[513-86-0]<br>ROWKJAVDOGWPAT-UHFFFAOYSA-N | 3.1<br>$5.7\times10^{-1}$<br>7.8<br>$1.1\times10^2$<br>$1.2\times10^1$<br>$9.9\times10^{-1}$ | | Wu et al. (2022b)<br>Straver and de Loos (2005)<br>Wang et al. (2017)<br>Wang et al. (2017)<br>Wang et al. (2017)<br>HSDB (2015) | M<br>M<br>Q<br>Q<br>Q<br>Q | 544<br><br>80, 238<br>80, 239<br>80, 240<br>99 |
| 2-(vinyloxy)ethanol<br>$C_4H_8O_2$<br>(ethylene glycol monovinyl ether)<br>[764-48-7]<br>VUIWJRYTWUGOOF-UHFFFAOYSA-N | $3.9\times10^1$ | | HSDB (2015) | Q | 99 |
| 2-methyloxetane<br>$C_4H_8O$<br>[2167-39-7]<br>FZIIBDOXPQOKBP-UHFFFAOYSA-N | $1.2\times10^{-1}$ | | HSDB (2015) | Q | 99 |
| 5-oxopentanoic acid<br>$OHC(CH_2)_3COOH$<br>[5746-02-1]<br>VBKPPDYGFUZOAJ-UHFFFAOYSA-N | $3.9\times10^1$ | | Saxena and Hildemann (1996) | E | 401 |
| 2-oxopentanedioic acid<br>$HOOC(CH_2)_2COCOOH$<br>($\alpha$-keto glutaric acid)<br>[328-50-7]<br>KPGXRSRHYNQIFN-UHFFFAOYSA-N | $9.9\times10^6$ | | Saxena and Hildemann (1996) | E | 401 |
| tetrahydro-2-furanmethanol<br>$C_5H_{10}O_2$<br>(tetrahydrofurfuryl alcohol)<br>[97-99-4]<br>BSYVTEYKTMYBMK-UHFFFAOYSA-N | $4.4\times10^1$<br>$1.7\times10^2$<br>$2.4\times10^3$ | | Yaws (2003)<br>Dupeux et al. (2022)<br>HSDB (2015) | X<br>Q<br>Q | 258<br>259<br>99 |



Table A3.12: Miscellaneous (... continued)

| Substance Formula (Trivial Name) [CAS Registry Number] InChIKey | $H_s^{cp}$ (at $T^\ominus$) $\left[\dfrac{\text{mol}}{\text{m}^3\,\text{Pa}}\right]$ | $\dfrac{\text{d}\ln H_s^{cp}}{\text{d}(1/T)}$ [K] | Reference | Type | Note |
|---|---|---|---|---|---|
| xylose $C_5H_{10}O_5$ [58-86-6] PYMYPHUHKUWMLA-VPENINKCSA-N | $8.2\times10^3$ | | HSDB (2015) | Q | 99 |
| 2-(2-methoxyethoxy)ethanol $C_5H_{12}O_3$ (diethylene glycol monomethyl ether) [111-77-3] SBASXUCJHJRPEV-UHFFFAOYSA-N | $6.2\times10^5$ | | HSDB (2015) | Q | 99 |
| 3,6-dioxaoctane-1,8-diol $HO(CH_2CH_2O)_3H$ (triethylene glycol) [112-27-6] ZIBGPFATKBEMQZ-UHFFFAOYSA-N | $1.1\times10^4$ $5.9\times10^3$ $6.7\times10^3$ $3.1\times10^5$ $8.9\times10^9$ | | Olsen et al. (2016) Olsen et al. (2016) Olsen et al. (2016) HSDB (2015) Saxena and Hildemann (1996) | Q Q Q Q E | 425 426 427 99 401 |
| 2-oxepanone $C_6H_{10}O_2$ (caprolactone) [502-44-3] PAPBSGBWRJIAAV-UHFFFAOYSA-N | $5.5\times10^{-2}$ | | HSDB (2015) | Q | 99 |
| glycidyl ether $C_6H_{10}O_3$ (diglycidyl ether) [2238-07-5] GYZLOYUZLJXAJU-UHFFFAOYSA-N | $7.6\times10^2$ | | HSDB (2015) | Q | 99 |
| 4-hydroxy-4-methyl-2-pentanone $C_6H_{12}O_2$ (diacetone alcohol) [123-42-2] SWXVUIWOUIDPGS-UHFFFAOYSA-N | $3.8\times10^1$ $1.3\times10^1$ $3.1\times10^2$ $7.4\times10^{-2}$ $7.6\times10^1$ $8.0\times10^2$ $6.8\times10^1$ $2.3\times10^3$ | | Duchowicz et al. (2020) Yaws (2003) Dupeux et al. (2022) Duchowicz et al. (2020) Wang et al. (2017) Wang et al. (2017) Wang et al. (2017) HSDB (2015) | V X Q Q Q Q Q Q | 186 258 259  80, 238 80, 239 80, 240 545 |
| 1-propoxy-2-propanol $C_6H_{14}O_2$ [1569-01-3] FENFUOGYJVOCRY-UHFFFAOYSA-N | $2.9\times10^2$ | | HSDB (2015) | Q | 545 |
| 2-(2-ethoxyethoxy)ethanol $C_6H_{14}O_3$ (diethylene glycol monoethyl ether) [111-90-0] XXJWXESWEXIICW-UHFFFAOYSA-N | $4.4\times10^2$ $4.5\times10^2$ $9.0\times10^2$ $6.4\times10^1$ | | Duchowicz et al. (2020) HSDB (2015) Dupeux et al. (2022) Duchowicz et al. (2020) | V V Q Q | 186  259 |





Table A3.12: Miscellaneous (... continued)

| Substance Formula (Trivial Name) [CAS Registry Number] InChIKey | $H_s^{cp}$ (at $T^{\ominus}$) $\left[\dfrac{\text{mol}}{\text{m}^3\,\text{Pa}}\right]$ | $\dfrac{\text{d}\ln H_s^{cp}}{\text{d}(1/T)}$ [K] | Reference | Type | Note |
|---|---|---|---|---|---|
| 2,5,8-trioxanonane $C_6H_{14}O_3$ (diglyme) [111-96-6] SBZXBUIDTXKZTM-UHFFFAOYSA-N | $1.9\times10^1$ $1.9\times10^1$ $6.2$ | | Duchowicz et al. (2020) HSDB (2015) Duchowicz et al. (2020) | V V Q | 186 |
| oxydipropanol $C_6H_{14}O_3$ (dipropylene glycol) [25265-71-8] SZXQTJUDPRGNJN-UHFFFAOYSA-N | $1.8\times10^3$ | | HSDB (2015) | V | |
| $p$-benzoquinone $C_6H_4O_2$ (1,4-benzoquinone) [106-51-4] AZQWKYJCGOJGHM-UHFFFAOYSA-N | $2.1\times10^{-2}$ $5.1$ $1.7\times10^4$ $2.3\times10^1$ $2.1\times10^{-2}$ $7.7$ | | HSDB (2015) Wang et al. (2017) Wang et al. (2017) Wang et al. (2017) Yaffe et al. (2003) Katritzky et al. (1998) | V Q Q Q Q Q | 80, 238 80, 239 80, 240 248, 249 |
| 5-hydroxymethylfurfural $C_6H_6O_3$ (5-hydroxymethyl-2-furfuraldehyde) [67-47-0] NOEGNKMFWQHSLB-UHFFFAOYSA-N | $1.8\times10^4$ | | HSDB (2015) | Q | 99 |
| 5-hydroxy-2-(hydroxymethyl)-4H-pyran-4-one $C_6H_6O_4$ (kojic acid) [501-30-4] BEJNERDRQOWKJM-UHFFFAOYSA-N | $4.1\times10^1$ | | HSDB (2015) | Q | 99 |
| 2-hydroxy-1,2,3-propanetricarboxylic acid $C_6H_8O_7$ (citric acid) [77-92-9] KRKNYBCHXYNGOX-UHFFFAOYSA-N | $3.1\times10^{15}$ $3.1\times10^{15}$ $7.6\times10^{12}$ $4.8\times10^{11}$ $7.9\times10^{12}$ $3.0\times10^{16}$ | | Burkholder et al. (2019) Burkholder et al. (2015) Compernolle and Müller (2014a) Yaws (2003) Gharagheizi et al. (2012) Gharagheizi et al. (2010) Saxena and Hildemann (1996) | L L V X Q Q E | 546 237, 12 246 401 |
| (butoxymethyl)oxirane $C_7H_{14}O_2$ (n-butyl glycidyl ether) [2426-08-6] YSUQLAYJZDEMOT-UHFFFAOYSA-N | $4.0\times10^{-1}$ $3.9\times10^{-1}$ $8.9\times10^{-1}$ | | Duchowicz et al. (2020) HSDB (2015) Duchowicz et al. (2020) | V V Q | 186 |



Table A3.12: Miscellaneous (...continued)

| Substance<br>Formula<br>(Trivial Name)<br>[CAS Registry Number]<br>InChIKey | $H_s^{cp}$<br>(at $T^{\ominus}$)<br><br>$\left[\dfrac{\text{mol}}{\text{m}^3\,\text{Pa}}\right]$ | $\dfrac{\text{d}\ln H_s^{cp}}{\text{d}(1/T)}$<br><br>[K] | Reference | Type | Note |
|---|---|---|---|---|---|
| 1-(1,1-dimethylethoxy)-2-propanol<br>$C_7H_{16}O_2$<br>(propylene glycol mono-t-butyl ether)<br>[57018-52-7]<br>GQCZPFJGIXHZMB-UHFFFAOYSA-N | 2.1 | | HSDB (2015) | V | |
| 2-[2-(2-methoxyethoxy)ethoxy]ethanol<br>$C_7H_{16}O_4$<br>(triethylene glycol monomethyl ether)<br>[112-35-6]<br>JLGLQAWTXXGVEM-UHFFFAOYSA-N | $2.8\times10^8$ | | HSDB (2015) | Q | 99 |
| 2-methyl-$p$-benzoquinone<br>$C_7H_6O_2$<br>[553-97-9]<br>VTWDKFNVVLAELH-UHFFFAOYSA-N | 3.1<br>$6.9\times10^3$<br>$2.1\times10^1$<br>$5.2\times10^3$ | | Wang et al. (2017)<br>Wang et al. (2017)<br>Wang et al. (2017)<br>HSDB (2015) | Q<br>Q<br>Q<br>Q | 80, 238<br>80, 239<br>80, 240<br>99 |
| patulin<br>$C_7H_6O_4$<br>[149-29-1]<br>ZRWPUFFVAOMMNM-UHFFFAOYSA-N | $9.0\times10^4$ | | HSDB (2015) | Q | 99 |
| 1-hydroxy-3-methoxybenzene<br>$C_7H_8O_2$<br>(3-methoxyphenol)<br>[150-19-6]<br>ASHGTJPOSUFTGB-UHFFFAOYSA-N | $1.7\times10^2$<br>$1.7\times10^2$<br>$4.7\times10^2$<br>$2.0\times10^2$<br>$2.5\times10^2$<br>$3.1\times10^2$<br>$1.3\times10^2$<br>$6.4\times10^1$<br>$5.0\times10^2$<br>$1.7\times10^2$ | | Abraham et al. (1994a)<br>Keshavarz et al. (2022)<br>Duchowicz et al. (2020)<br>Raventos-Duran et al. (2010)<br>Raventos-Duran et al. (2010)<br>Raventos-Duran et al. (2010)<br>Hilal et al. (2008)<br>Modarresi et al. (2007)<br>Nirmalakhandan et al. (1997)<br>Duchowicz et al. (2020) | R<br>Q<br>Q<br>Q<br>Q<br>Q<br>Q<br>Q<br>Q<br>? | <br><br>299<br>242, 243<br>244<br>245<br><br>67<br><br>185, 21 |
| mequinol<br>$C_7H_8O_2$<br>(4-methoxyphenol)<br>[150-76-5]<br>NWVVVBRKAWDGAB-UHFFFAOYSA-N | $6.9\times10^{-1}$<br>$1.9\times10^1$<br>$6.9\times10^{-1}$<br>2.0 | | Yaws (2003)<br>HSDB (2015)<br>Gharagheizi et al. (2010)<br>Yaws (1999) | X<br>Q<br>Q<br>? | 237, 14<br>99<br>246<br>21, 14 |
| 1-hydroxy-2-methoxybenzene<br>$C_7H_8O_2$<br>(guaiacol; 2-methoxyphenol)<br>[90-05-1]<br>LHGVFZTZFXWLCP-UHFFFAOYSA-N | 8.6<br>9.0<br>7.7<br>9.1<br>9.6<br>7.7<br>$4.1\times10^1$<br>5.0 | <br>7900<br><br>7600 | McFall et al. (2020)<br>Wieland et al. (2015)<br>Sagebiel et al. (1992)<br>Sagebiel et al. (1992)<br>Mackay et al. (2006c)<br>Sagebiel et al. (1992)<br>Leuenberger et al. (1985)<br>Abraham et al. (1994a) | M<br>M<br>M<br>M<br>V<br>V<br>V<br>R | <br>547<br><br><br><br><br>416<br> |



Table A3.12: Miscellaneous (. . . continued)

| Substance Formula (Trivial Name) [CAS Registry Number] InChIKey | $H_s^{cp}$ (at $T^\ominus$) $\left[\dfrac{\text{mol}}{\text{m}^3\,\text{Pa}}\right]$ | $\dfrac{\text{d}\ln H_s^{cp}}{\text{d}(1/T)}$ [K] | Reference | Type | Note |
|---|---|---|---|---|---|
| | $1.1\times10^1$ | | Keshavarz et al. (2022) | Q | |
| | $5.6\times10^{-1}$ | | Abney (2021) | Q | 399 |
| | 8.7 | | Duchowicz et al. (2020) | Q | 299 |
| | 8.2 | | McFall et al. (2020) | Q | 474 |
| | 5.2 | | Hilal et al. (2008) | Q | |
| | 7.7 | | Modarresi et al. (2007) | Q | 67 |
| | | 6700 | Kühne et al. (2005) | Q | |
| | $1.2\times10^1$ | | English and Carroll (2001) | Q | 230, 274 |
| | $6.4\times10^1$ | | Katritzky et al. (1998) | Q | |
| | $5.1\times10^2$ | | Nirmalakhandan et al. (1997) | Q | |
| | 8.2 | | Duchowicz et al. (2020) | ? | 185, 21 |
| | | 7800 | Kühne et al. (2005) | ? | |
| | $3.3\times10^{-1}$ | | Yaws (1999) | ? | 21, 14 |
| 1,4-dimethoxybenzene $C_8H_{10}O_2$ (hydroquinone dimethyl ether) [150-78-7] OHBQPCCCRFSCAX-UHFFFAOYSA-N | $2.8\times10^{-3}$ | | HSDB (2015) | Q | 99 |
| 4-methyl-2-methoxyphenol $C_8H_{10}O_2$ [93-51-6] PETRWTHZSKVLRE-UHFFFAOYSA-N | 7.7 | | Sagebiel et al. (1992) | M | |
| | 7.1 | 7400 | Sagebiel et al. (1992) | M | |
| | $1.0\times10^1$ | | Sagebiel et al. (1992) | V | |
| | $1.6\times10^1$ | | Keshavarz et al. (2022) | Q | |
| | 5.2 | | Duchowicz et al. (2020) | Q | |
| | $3.1\times10^1$ | | Raventos-Duran et al. (2010) | Q | 271, 243 |
| | 3.1 | | Raventos-Duran et al. (2010) | Q | 244 |
| | $2.5\times10^2$ | | Raventos-Duran et al. (2010) | Q | 245 |
| | 5.2 | | Hilal et al. (2008) | Q | |
| | 2.5 | | Modarresi et al. (2007) | Q | 67 |
| | | 7100 | Kühne et al. (2005) | Q | |
| | 7.3 | | Yaffe et al. (2003) | Q | 248, 249 |
| | $6.2\times10^1$ | | Katritzky et al. (1998) | Q | |
| | 7.4 | | Duchowicz et al. (2020) | ? | 185, 21 |
| | | 7900 | Kühne et al. (2005) | ? | |
| tyrosol $C_8H_{10}O_2$ (4-hydroxybenzeneethanol) [501-94-0] YCCILVSKPBXVIP-UHFFFAOYSA-N | $3.5\times10^5$ | | McFall et al. (2020) | Q | 474 |
| 1,3-dimethoxy-2-hydroxybenzene $C_8H_{10}O_3$ (2,6-dimethoxyphenol) [91-10-1] KLIDCXVFHGNTTM-UHFFFAOYSA-N | $3.7\times10^1$ | | Sagebiel et al. (1992) | M | |
| | $5.0\times10^1$ | 6700 | Sagebiel et al. (1992) | M | |
| | $1.2\times10^2$ | | Sagebiel et al. (1992) | V | |
| | $1.6\times10^1$ | | Keshavarz et al. (2022) | Q | |
| | 1.2 | | Duchowicz et al. (2020) | Q | |
| | $7.8\times10^2$ | | Raventos-Duran et al. (2010) | Q | 271, 243 |
| | $4.9\times10^2$ | | Raventos-Duran et al. (2010) | Q | 244 |



Table A3.12: Miscellaneous (... continued)

| Substance Formula (Trivial Name) [CAS Registry Number] InChIKey | $H_s^{cp}$ (at $T^\ominus$) $\left[\dfrac{\mathrm{mol}}{\mathrm{m^3\,Pa}}\right]$ | $\dfrac{\mathrm{d}\ln H_s^{cp}}{\mathrm{d}(1/T)}$ [K] | Reference | Type | Note |
|---|---|---|---|---|---|
| | $4.9\times10^3$ | | Raventos-Duran et al. (2010) | Q | 245 |
| | $3.5\times10^2$ | | Hilal et al. (2008) | Q | |
| | $1.1\times10^2$ | | Modarresi et al. (2007) | Q | 67 |
| | | 7300 | Kühne et al. (2005) | Q | |
| | $4.3\times10^1$ | | Duchowicz et al. (2020) | ? | 185, 21 |
| | | 7600 | Kühne et al. (2005) | ? | |
| hexahydro-1,3-isobenzofurandione $C_8H_{10}O_3$ (hexahydrophthalic anhydride) [85-42-7] MUTGBJKUEZFXGO-UHFFFAOYSA-N | $4.7\times10^{-1}$ | | HSDB (2015) | Q | 99 |
| vanillyl alcohol $C_8H_{10}O_3$ [498-00-0] ZENOXNGFMSCLLL-UHFFFAOYSA-N | $9.6\times10^4$ $2.0\times10^5$ | | McFall et al. (2020) McFall et al. (2020) | M Q | 474 |
| 1-methoxy-4-methylbenzene $C_8H_{10}O$ [104-93-8] CHLICZRVGGXEOD-UHFFFAOYSA-N | $2.1\times10^{-3}$ | | HSDB (2015) | Q | 99 |
| dimethoxane $C_8H_{14}O_4$ [828-00-2] PHMNXPYGVPEQSJ-UHFFFAOYSA-N | $8.0\times10^1$ | | HSDB (2015) | Q | 99 |
| RO5R5 $C_8H_{14}O_4$ MESHXYGXEHAFFE-UHFFFAOYSA-N | $1.3\times10^5$ | 17000 | Wieser et al. (2023) | Q | 437 |
| 5-hydroxy-2-octanone $C_8H_{16}O_2$ (C82CO5OH) KZPPEBIAPHLFQD-UHFFFAOYSA-N | $1.6\times10^3$ | 11000 | Wieser et al. (2023) | Q | 437 |
| metaldehyde $C_8H_{16}O_4$ [108-62-3] GKKDCARASOJPNG-UHFFFAOYSA-N | $1.9\times10^{-1}$ $2.9\times10^{-1}$ | | HSDB (2015) Maniere et al. (2011) | V ? | 12, 165 |
| diethyl carbitol $C_8H_{18}O_3$ (diethylene glycol diethyl ether) [112-36-7] RRQYJINTUHWNHW-UHFFFAOYSA-N | $8.9\times10^1$ $9.0\times10^1$ $9.6\times10^{-1}$ | | Duchowicz et al. (2020) HSDB (2015) Duchowicz et al. (2020) | V V Q | 186 |





Table A3.12: Miscellaneous (. . . continued)

| Substance Formula (Trivial Name) [CAS Registry Number] InChIKey | $H_s^{cp}$ (at $T^\ominus$) $\left[\dfrac{\text{mol}}{\text{m}^3\,\text{Pa}}\right]$ | $\dfrac{\text{d}\ln H_s^{cp}}{\text{d}(1/T)}$ [K] | Reference | Type | Note |
|---|---|---|---|---|---|
| 2-[2-(2-ethoxyethoxy)ethoxy]ethanol $C_8H_{18}O_4$ (triethylene glycol monoethyl ether) [112-50-5] WFSMVVDJSNMRAR-UHFFFAOYSA-N | $2.1\times10^8$ | | HSDB (2015) | Q | 99 |
| tetraethylene glycol $C_8H_{18}O_5$ [112-60-7] UWHCKJMYHZGTIT-UHFFFAOYSA-N | $1.8\times10^{13}$ | | HSDB (2015) | Q | 99 |
| vanillin $C_8H_8O_3$ [121-33-5] MWOOGOJBHIARFG-UHFFFAOYSA-N | $1.7\times10^3$ | 6800 | Brockbank (2013) | L | 1 |
| | $4.6\times10^3$ | | Duchowicz et al. (2020) | V | 186 |
| | $4.7\times10^3$ | | HSDB (2015) | V | |
| | $1.3\times10^3$ | | Yaws (2003) | X | 258 |
| | $1.3\times10^3$ | | Yaws (2003) | X | 237 |
| | $1.8\times10^3$ | | Dupeux et al. (2022) | Q | 259 |
| | $1.8\times10^3$ | | Dupeux et al. (2022) | Q | 259 |
| | $1.8\times10^3$ | | Abney (2021) | Q | 399 |
| | $4.0\times10^3$ | | Duchowicz et al. (2020) | Q | |
| | $2.8\times10^2$ | | Gharagheizi et al. (2012) | Q | |
| | $4.9\times10^3$ | | Raventos-Duran et al. (2010) | Q | 242, 243 |
| | $3.1\times10^2$ | | Raventos-Duran et al. (2010) | Q | 244 |
| | $1.2\times10^5$ | | Raventos-Duran et al. (2010) | Q | 245 |
| | $1.3\times10^3$ | | Gharagheizi et al. (2010) | Q | 246 |
| ethylparaben $C_9H_{10}O_3$ [120-47-8] NUVBSKCKDOMJSU-UHFFFAOYSA-N | $2.1\times10^3$ | | HSDB (2015) | Q | 99 |
| ethyl vanillin $C_9H_{10}O_3$ [121-32-4] CBOQJANXLMLOSS-UHFFFAOYSA-N | $1.2\times10^4$ | | HSDB (2015) | V | |
| 1-phenoxypropan-2-ol $C_9H_{12}O_2$ (propylene glycol phenyl ether) [770-35-4] IBLKWZIFZMJLFL-UHFFFAOYSA-N | $3.4\times10^2$ | | HSDB (2015) | V | |
| triacetin $C_9H_{14}O_6$ [102-76-1] URAYPUMNDPQOKB-UHFFFAOYSA-N | $8.0\times10^2$ | | Duchowicz et al. (2020) | V | 186 |
| | $8.2\times10^2$ | | HSDB (2015) | V | |
| | $2.7\times10^2$ | | Duchowicz et al. (2020) | Q | |
| | $2.0\times10^3$ | | Raventos-Duran et al. (2010) | Q | 242, 243 |
| | $2.5\times10^2$ | | Raventos-Duran et al. (2010) | Q | 244 |
| | $6.2\times10^3$ | | Raventos-Duran et al. (2010) | Q | 245 |



Table A3.12: Miscellaneous (...continued)

| Substance Formula (Trivial Name) [CAS Registry Number] InChIKey | $H_s^{cp}$ (at $T^{\ominus}$) $\left[\dfrac{\mathrm{mol}}{\mathrm{m^3\,Pa}}\right]$ | $\dfrac{\mathrm{d}\ln H_s^{cp}}{\mathrm{d}(1/T)}$ [K] | Reference | Type | Note |
|---|---|---|---|---|---|
| tripropylene glycol $C_9H_{20}O_4$ [24800-44-0] LCZVSXRMYJUNFX-UHFFFAOYSA-N | $3.0 \times 10^9$ | | HSDB (2015) | Q | 99 |
| coumarin $C_9H_6O_2$ [91-64-5] ZYGHJZDHTFUPRJ-UHFFFAOYSA-N | $9.9 \times 10^1$ $1.0 \times 10^2$ $1.3 \times 10^2$ 5.1 | | Duchowicz et al. (2020) HSDB (2015) Dupeux et al. (2022) Duchowicz et al. (2020) | V V Q Q | 186 259 |
| 4-ethylguaiacol $C_9H_{12}O_2$ (4-ethyl-2-methoxyphenol) [2785-89-9] CHWNEIVBYREQRF-UHFFFAOYSA-N | 5.5 | | McFall et al. (2020) | Q | 474 |
| 4-methylsyringol $C_9H_{12}O_3$ [6638-05-7] ZFBNNSOJNZBLLS-UHFFFAOYSA-N | $3.8 \times 10^1$ | | McFall et al. (2020) | Q | 474 |
| acetovanillone $C_9H_{10}O_3$ (apocynin) [498-02-2] DFYRUELUNQRZTB-UHFFFAOYSA-N | $6.3 \times 10^3$ | | McFall et al. (2020) | Q | 474 |
| syringaldehyde $C_9H_{10}O_4$ [134-96-3] KCDXJAYRVLXPFO-UHFFFAOYSA-N | $3.7 \times 10^5$ | | McFall et al. (2020) | Q | 474 |
| 5-(1-propenyl)-1,3-benzodioxole $C_{10}H_{10}O_2$ (isosafrole) [120-58-1] VHVOLFRBFDOUSH-UHFFFAOYSA-N | $2.7 \times 10^{-4}$ | | HSDB (2015) | Q | 99 |
| safrole $C_{10}H_{10}O_2$ [94-59-7] ZMQAAUBTXCXRIC-UHFFFAOYSA-N | 1.1 | | HSDB (2015) | Q | 99 |
| 5-propyl-1,3-benzodioxole $C_{10}H_{12}O_2$ (dihydrosafrole) [94-58-6] MYEIDJPOUKASEC-UHFFFAOYSA-N | $8.2 \times 10^{-1}$ | | HSDB (2015) | Q | 99 |





Table A3.12: Miscellaneous (...continued)

| Substance Formula (Trivial Name) [CAS Registry Number] InChIKey | $H_s^{cp}$ (at $T^\ominus$) $\left[\dfrac{\mathrm{mol}}{\mathrm{m^3\,Pa}}\right]$ | $\dfrac{\mathrm{d\ln}H_s^{cp}}{\mathrm{d}(1/T)}$ [K] | Reference | Type | Note |
|---|---|---|---|---|---|
| 2-methoxy-4-(1-propenyl)phenol $C_{10}H_{12}O_2$ (isoeugenol) [97-54-1] BJIOGJUNALELMI-UHFFFAOYSA-N | 2.7 $1.7\times10^1$ | | HSDB (2015) McFall et al. (2020) | V Q | 474 |
| $p$-cresyl glycidyl ether $C_{10}H_{12}O_2$ [26447-14-3] CUFXMPWHOWYNSO-UHFFFAOYSA-N | $1.3\times10^1$ | | HSDB (2015) | Q | 99 |
| 4-(4-hydroxyphenyl)-2-butanone $C_{10}H_{12}O_2$ (raspberry ketone) [5471-51-2] NJGBTKGETPDVIK-UHFFFAOYSA-N | $1.8\times10^4$ | | HSDB (2015) | Q | 447 |
| guaifenesin $C_{10}H_{14}O_4$ [93-14-1] HSRJKNPTNIJEKV-UHFFFAOYSA-N | $2.2\times10^5$ | | HSDB (2015) | Q | 99 |
| RO5R4O2H $C_{10}H_{18}O_7$ YORZXZYHORHTRW-UHFFFAOYSA-N | $3.6\times10^{10}$ | 24000 | Wieser et al. (2023) | Q | 437 |
| levomenthol $C_{10}H_{20}O$ ($L$-menthol) [2216-51-5] NOOLISFMXDJSKH-AEJSXWLSSA-N | $6.6\times10^{-1}$ | | HSDB (2015) | Q | 99 |
| diethylene glycol hexyl ether $C_{10}H_{22}O_3$ [112-59-4] GZMAAYIALGURDQ-UHFFFAOYSA-N | $5.7\times10^2$ $5.8\times10^2$ $1.0\times10^2$ | | Duchowicz et al. (2020) HSDB (2015) Duchowicz et al. (2020) | V V Q | 186 |
| 2-[2-(2-butoxyethoxy)ethoxy]ethanol $C_{10}H_{22}O_4$ (triethylene glycol monobutyl ether) [143-22-6] COBPKKZHLDDMTB-UHFFFAOYSA-N | $1.0\times10^8$ | | HSDB (2015) | Q | 99 |
| 4-propylguaiacol $C_{10}H_{14}O_2$ (2-methoxy-4-propylphenol) [2785-87-7] PXIKRTCSSLJURC-UHFFFAOYSA-N | 4.3 4.2 | | McFall et al. (2020) McFall et al. (2020) | M Q | 474 |





Table A3.12: Miscellaneous (...continued)

| Substance Formula (Trivial Name) [CAS Registry Number] InChIKey | $H_s^{cp}$ (at $T^\ominus$) $\left[\dfrac{\text{mol}}{\text{m}^3\,\text{Pa}}\right]$ | $\dfrac{\text{d}\ln H_s^{cp}}{\text{d}(1/T)}$ [K] | Reference | Type | Note |
|---|---|---|---|---|---|
| vanillyl ethyl ether $C_{10}H_{14}O_3$ (4-(ethoxymethyl)-2-methoxyphenol) [13184-86-6] KOCVACNWDMSLBM-UHFFFAOYSA-N | $4.1\times10^2$ $4.8\times10^2$ | | McFall et al. (2020) McFall et al. (2020) | M Q | 474 |
| 4-ethylsyringol $C_{10}H_{14}O_3$ [14059-92-8] PJWDIHUFLXQRFF-UHFFFAOYSA-N | $3.0\times10^1$ | | McFall et al. (2020) | Q | 474 |
| guaiacylacetone $C_{10}H_{12}O_3$ (1-(4-hydroxy-3-methoxyphenyl)-2-propanone) [2503-46-0] LFVCJQWZGDLHSD-UHFFFAOYSA-N | $1.2\times10^4$ $1.1\times10^4$ | | McFall et al. (2020) McFall et al. (2020) | M Q | 474 |
| acetosyringone $C_{10}H_{12}O_4$ [2478-38-8] OJOBTAOGJIWAGB-UHFFFAOYSA-N | $5.0\times10^5$ | | McFall et al. (2020) | Q | 474 |
| coniferylaldehyde $C_{10}H_{10}O_3$ [458-36-6] UCZOBXQKIUMJDZ-UHFFFAOYSA-N | $4.4\times10^4$ | | McFall et al. (2020) | Q | 474 |
| 4-methoxy-6-(2-propenyl)-1,3-benzodioxole $C_{11}H_{12}O_3$ (myristicin) [607-91-0] BNWJOHGLIBDBOB-UHFFFAOYSA-N | $1.8\times10^1$ | | HSDB (2015) | Q | 99 |
| butylparaben $C_{11}H_{14}O_3$ [94-26-8] QFOHBWFCKVYLES-UHFFFAOYSA-N | $1.2\times10^3$ | | HSDB (2015) | Q | 99 |
| 2-*tert*-butyl-4-methoxyphenol $C_{11}H_{16}O_2$ (butylated hydroxyanisole) [25013-16-5] MRBKEAMVRSLQPH-UHFFFAOYSA-N | 8.4 | | HSDB (2015) | Q | 99 |





Table A3.12: Miscellaneous (...continued)

| Substance Formula (Trivial Name) [CAS Registry Number] InChIKey | $H_s^{cp}$ (at $T^\ominus$) $\left[\dfrac{\text{mol}}{\text{m}^3\,\text{Pa}}\right]$ | $\dfrac{\text{d}\ln H_s^{cp}}{\text{d}(1/T)}$ [K] | Reference | Type | Note |
|---|---|---|---|---|---|
| 3-hydroxy-2-naphthalenecarboxylic acid | $7.0 \times 10^3$ | | HSDB (2015) | Q | 99 |
| $C_{11}H_8O_3$ | $7.2 \times 10^3$ | | Zhang et al. (2010) | Q | 287, 288 |
| [92-70-6] | $1.2 \times 10^4$ | | Zhang et al. (2010) | Q | 287, 289 |
| ALKYHXVLJMQRLQ-UHFFFAOYSA-N | $3.8 \times 10^5$ | | Zhang et al. (2010) | Q | 287, 290 |
| | $8.2 \times 10^3$ | | Zhang et al. (2010) | Q | 287, 291 |
| 4-propylsyringol $C_{11}H_{16}O_3$ [6766-82-1] YHEWWEXPVKCVFY-UHFFFAOYSA-N | $2.2 \times 10^1$ | | McFall et al. (2020) | Q | 474 |
| zingerone | $1.2 \times 10^4$ | | McFall et al. (2020) | M | |
| $C_{11}H_{14}O_3$ | $8.4 \times 10^3$ | | McFall et al. (2020) | Q | 474 |
| (vanillylacetone) [122-48-5] OJYLAHXKWMRDGS-UHFFFAOYSA-N | | | | | |
| allylsyringol $C_{11}H_{14}O_3$ [5438-54-0] IYIVYBWBUQPWAU-UHFFFAOYSA-N | $6.3 \times 10^2$ | | McFall et al. (2020) | Q | 474 |
| 4-propenylsyringol $C_{11}H_{14}O_3$ [6635-22-9] YFHOHYAUMDHSBX-UHFFFAOYSA-N | $1.1 \times 10^3$ | | McFall et al. (2020) | Q | 474 |
| propionylsyringol $C_{11}H_{14}O_4$ [5650-43-1] CXCPJZXJNRBTGF-UHFFFAOYSA-N | $3.8 \times 10^5$ | | McFall et al. (2020) | Q | 474 |
| sinapylaldehyde $C_{11}H_{12}O_4$ [4206-58-0] CDICDSOGTRCHMG-ONEGZZNKSA-N | $3.1 \times 10^6$ | | McFall et al. (2020) | Q | 474 |
| arbutin $C_{12}H_{16}O_7$ [497-76-7] BJRNKVDFDLYUGJ-RMPHRYRLSA-N | $8.2 \times 10^{13}$ | | HSDB (2015) | Q | 99 |
| butopyronoxyl $C_{12}H_{18}O_4$ (indalone) [532-34-3] OKIJSNGRQAOIGZ-UHFFFAOYSA-N | $2.1 \times 10^2$ | | HSDB (2015) | Q | 99 |





Table A3.12: Miscellaneous (...continued)

| Substance Formula (Trivial Name) [CAS Registry Number] InChIKey | $H_s^{cp}$ (at $T^\ominus$) $\left[\dfrac{\mathrm{mol}}{\mathrm{m^3\,Pa}}\right]$ | $\dfrac{\mathrm{d}\ln H_s^{cp}}{\mathrm{d}(1/T)}$ [K] | Reference | Type | Note |
|---|---|---|---|---|---|
| diethylene glycol bis(methacrylate) $C_{12}H_{18}O_5$ [2358-84-1] XFCMNSHQOZQILR-UHFFFAOYSA-N | $1.2 \times 10^4$ | | HSDB (2015) | Q | 99 |
| dikegulac $C_{12}H_{18}O_7$ [18467-77-1] FWCBATIDXGJRMF-FLNNQWSLSA-N | $5.2 \times 10^{10}$ | | HSDB (2015) | Q | 99 |
| propofol $C_{12}H_{18}O$ [2078-54-8] OLBCVFGFOZPWHH-UHFFFAOYSA-N | $4.7$ | | HSDB (2015) | Q | 99 |
| lactitol $C_{12}H_{24}O_{11}$ [585-86-4] VQHSOMBJVWLPSR-JVCRWLNRSA-N | $1.2 \times 10^{16}$ | | HSDB (2015) | Q | 99 |
| maltitol $C_{12}H_{24}O_{11}$ [585-88-6] VQHSOMBJVWLPSR-WUJBLJFYSA-N | $2.3 \times 10^{15}$ | | HSDB (2015) | Q | 99 |
| naphthalic anhydride $C_{12}H_6O_3$ [81-84-5] GRSMWKLPSNHDHA-UHFFFAOYSA-N | $1.6 \times 10^1$ | | HSDB (2015) | Q | 99 |
| methoxsalen $C_{12}H_8O_4$ (8-methoxypsoralen) [298-81-7] QXKHYNVANLEOEG-UHFFFAOYSA-N | $2.5 \times 10^2$ | | HSDB (2015) | Q | 99 |
| syringylacetone $C_{12}H_{16}O_4$ [112468-41-4] NULBEPOZYDYWOV-UHFFFAOYSA-N | $1.2 \times 10^6$ $5.7 \times 10^4$ | | McFall et al. (2020) McFall et al. (2020) | M Q | 474 |
| butyrylsyringol $C_{12}H_{16}O_4$ [69271-91-6] QFHXMVPZYYMTCS-UHFFFAOYSA-N | $2.9 \times 10^5$ | | McFall et al. (2020) | Q | 474 |
| bisphenol F $C_{13}H_{12}O_2$ [620-92-8] PXKLMJQFEQBVLD-UHFFFAOYSA-N | $1.9 \times 10^6$ | | HSDB (2015) | Q | 447 |



Table A3.12: Miscellaneous (. . . continued)

| Substance<br>Formula<br>(Trivial Name)<br>[CAS Registry Number]<br>InChIKey | $H_s^{cp}$<br>(at $T^\ominus$)<br>$\left[\dfrac{\mathrm{mol}}{\mathrm{m^3\,Pa}}\right]$ | $\dfrac{\mathrm{d}\ln H_s^{cp}}{\mathrm{d}(1/T)}$<br><br>[K] | Reference | Type | Note |
|---|---|---|---|---|---|
| ibuprofen<br>$C_{13}H_{18}O_2$<br>[15687-27-1]<br>HEFNNWSXXWATRW-UHFFFAOYSA-N | $6.6\times10^1$<br>$2.0\times10^2$ | | HSDB (2015)<br>Abraham et al. (2019) | V<br>Q | |
| trinexapac-ethyl<br>$C_{13}H_{16}O_5$<br>[95266-40-3]<br>RVKCCVTVZORVGD-QXMHVHEDSA-N | 5.1<br>$7.6\times10^5$<br>$1.9\times10^3$ | | Duchowicz et al. (2020)<br>Duchowicz et al. (2020)<br>Maniere et al. (2011) | V<br>Q<br>? | 186<br><br>165 |
| benzoyl peroxide<br>$C_{14}H_{10}O_4$<br>[94-36-0]<br>OMPJBNCRMGITSC-UHFFFAOYSA-N | 2.8<br>2.8<br>$1.1\times10^2$<br>$4.1\times10^2$<br>$4.3\times10^3$ | | HSDB (2015)<br>Zhang et al. (2010)<br>Zhang et al. (2010)<br>Zhang et al. (2010)<br>Zhang et al. (2010) | Q<br>Q<br>Q<br>Q<br>Q | 99<br>287, 288<br>287, 289<br>287, 290<br>287, 291 |
| oxybenzone<br>$C_{14}H_{12}O_3$<br>(2-hydroxy-4-<br>methoxybenzophenone)<br>[131-57-7]<br>DXGLGDHPHMLXJC-UHFFFAOYSA-N | $6.6\times10^2$ | | HSDB (2015) | Q | 99 |
| resveratrol<br>$C_{14}H_{12}O_3$<br>[501-36-0]<br>LUKBXSAWLPMMSZ-OWOJBTEDSA-N | $7.0\times10^{10}$ | | HSDB (2015) | Q | 447 |
| pindone<br>$C_{14}H_{14}O_3$<br>[83-26-1]<br>RZKYEQDPDZUERB-UHFFFAOYSA-N | $1.1\times10^6$ | | HSDB (2015) | Q | 99 |
| 1,1'-<br>[oxybis(methylene)]bisbenzene<br>$C_{14}H_{14}O$<br>(dibenzyl ether)<br>[103-50-4]<br>MHDVGSVTJDSBDK-UHFFFAOYSA-N | 1.5<br>$3.7\times10^{-1}$<br>$3.1\times10^{-1}$<br>$1.2\times10^2$<br>3.6 | | Duchowicz et al. (2020)<br>Dupeux et al. (2022)<br>Duchowicz et al. (2020)<br>HSDB (2015)<br>Modarresi et al. (2007) | V<br>Q<br>Q<br>Q<br>Q | 186<br>259<br><br>99<br>67 |
| butanoic acid,<br>3,3-bis((1,1-dimethylethyl)dioxy)-,<br>ethyl ester<br>$C_{14}H_{28}O_6$<br>[55794-20-2]<br>HARQWLDROVMFJE-UHFFFAOYSA-N | 5.0<br><br><br>$7.0\times10^{-3}$<br>$1.3\times10^2$<br>$2.4\times10^2$ | | Zhang et al. (2010)<br><br><br>Zhang et al. (2010)<br>Zhang et al. (2010)<br>Zhang et al. (2010) | Q<br><br><br>Q<br>Q<br>Q | 287, 288<br><br><br>287, 289<br>287, 290<br>287, 291 |




Table A3.12: Miscellaneous (...continued)

| Substance Formula (Trivial Name) [CAS Registry Number] InChIKey | $H_s^{cp}$ (at $T^\ominus$) $\left[\dfrac{\text{mol}}{\text{m}^3\,\text{Pa}}\right]$ | $\dfrac{\mathrm{d}\ln H_s^{cp}}{\mathrm{d}(1/T)}$ [K] | Reference | Type | Note |
|---|---|---|---|---|---|
| 1-hydroxy-9,10-anthracenedione $C_{14}H_8O_3$ (1-hydroxyanthraquinone) [129-43-1] BTLXPCBPYBNQNR-UHFFFAOYSA-N | $1.4\times10^3$ $1.4\times10^3$ $3.5\times10^3$ | | Duchowicz et al. (2020) HSDB (2015) Duchowicz et al. (2020) | V V Q | 186 |
| danthron $C_{14}H_8O_4$ (1,8-dihydroxyanthraquinone) [117-10-2] QBPFLULOKWLNNW-UHFFFAOYSA-N | $1.8\times10^5$ | | HSDB (2015) | Q | 99 |
| bisphenol A $C_{15}H_{16}O_2$ [80-05-7] IISBACLAFKSPIT-UHFFFAOYSA-N | $2.5\times10^5$ | | HSDB (2015) | V | |
| atractylenolide III $C_{15}H_{20}O_3$ [73030-71-4] FBMORZZOJSDNRQ-GLQYFDAESA-N | $1.0\times10^3$ | | HSDB (2015) | Q | 99 |
| deoxynivalenol $C_{15}H_{20}O_6$ [51481-10-8] LINOMUASTDIRTM-WHNKEALZSA-N | $4.9\times10^8$ | | HSDB (2015) | Q | 447 |
| nivalenol $C_{15}H_{20}O_7$ [23282-20-4] UKOTXHQERFPCBU-UHFFFAOYSA-N | $1.4\times10^{10}$ | | HSDB (2015) | Q | 99 |
| tributyrin $C_{15}H_{26}O_6$ [60-01-5] UYXTWWCETRIEDR-UHFFFAOYSA-N | $1.0\times10^3$ | | HSDB (2015) | Q | 99 |
| diosmetin $C_{16}H_{12}O_6$ [520-34-3] MBNGWHIJMBWFHU-UHFFFAOYSA-N | $3.3\times10^{12}$ | | HSDB (2015) | Q | 447 |
| shikonin $C_{16}H_{16}O_5$ [517-89-5] NEZONWMXZKDMKF-SNVBAGLBSA-N | $1.2\times10^9$ | | HSDB (2015) | Q | 447 |



Table A3.12: Miscellaneous (... continued)

| Substance Formula (Trivial Name) [CAS Registry Number] InChIKey | $H_s^{cp}$ (at $T^{\ominus}$) $\left[\dfrac{\mathrm{mol}}{\mathrm{m}^3\,\mathrm{Pa}}\right]$ | $\dfrac{\mathrm{d}\ln H_s^{cp}}{\mathrm{d}(1/T)}$ [K] | Reference | Type | Note |
|---|---|---|---|---|---|
| 2,2-bis(4-hydroxyphenyl)butane $C_{16}H_{18}O_2$ (bisphenol B) [77-40-7] HTVITOHKHWFJKO-UHFFFAOYSA-N | $8.2\times10^5$ | | HSDB (2015) | Q | 447 |
| ethyl 3,3-bis(*tert*-amylperoxy)butyrate $C_{16}H_{32}O_6$ [67567-23-1] NICWAKGKDIAMOD-UHFFFAOYSA-N | 2.9 $3.7\times10^{-3}$ $3.0\times10^1$ $1.5\times10^2$ | | Zhang et al. (2010) Zhang et al. (2010) Zhang et al. (2010) Zhang et al. (2010) | Q Q Q Q | 287, 288 287, 289 287, 290 287, 291 |
| aflatoxin B1 $C_{17}H_{12}O_6$ [1162-65-8] OQIQSTLJSLGHID-UHFFFAOYSA-N | $7.0\times10^7$ | | HSDB (2015) | Q | 99 |
| aflatoxin G1 $C_{17}H_{12}O_7$ [1165-39-5] XWIYFDMXXLINPU-UHFFFAOYSA-N | $2.0\times10^7$ | | HSDB (2015) | Q | 99 |
| aflatoxin B2 $C_{17}H_{14}O_6$ [7220-81-7] WWSYXEZEXMQWHT-UHFFFAOYSA-N | $3.3\times10^9$ | | HSDB (2015) | Q | 99 |
| aflatoxin G2 $C_{17}H_{14}O_7$ [7241-98-7] WPCVRWVBBXIRMA-UHFFFAOYSA-N | $9.0\times10^8$ | | HSDB (2015) | Q | 99 |
| bisphenol C $C_{17}H_{20}O_2$ [79-97-0] YMTYZTXUZLQUSF-UHFFFAOYSA-N | $9.0\times10^5$ | | HSDB (2015) | Q | 447 |
| PR-toxin $C_{17}H_{20}O_6$ [56299-00-4] GSPFUBNBRPVALJ-VIEAGMIOSA-N | $1.6\times10^8$ | | HSDB (2015) | Q | 99 |
| fusarenon X $C_{17}H_{22}O_8$ [23255-69-8] XGCUCFKWVIWWNW-UHFFFAOYSA-N | $2.1\times10^{11}$ | | HSDB (2015) | Q | 99 |
| dihydrotanshinone I $C_{18}H_{14}O_3$ [87205-99-0] HARGZZNYNSYSGJ-JTQLQIEISA-N | $7.6\times10^4$ | | HSDB (2015) | Q | 99 |



Table A3.12: Miscellaneous (. . . continued)

| Substance Formula (Trivial Name) [CAS Registry Number] InChIKey | $H_s^{cp}$ (at $T^\ominus$) $\left[\dfrac{\mathrm{mol}}{\mathrm{m^3\,Pa}}\right]$ | $\dfrac{\mathrm{d}\ln H_s^{cp}}{\mathrm{d}(1/T)}$ [K] | Reference | Type | Note |
|---|---|---|---|---|---|
| diethylstilbestrol $C_{18}H_{20}O_2$ [56-53-1] RGLYKWWBQGJZGM-ISLYRVAYSA-N | $1.7\times10^2$ | | HSDB (2015) | Q | 99 |
| laminarin $C_{18}H_{32}O_{16}$ [9008-22-4] DBTMGCOVALSLOR-AWHOAIGYSA-N | $>2.9\times10^6$ | | Maniere et al. (2011) | ? | 72, 165 |
| estrone $C_{18}H_{22}O_2$ [53-16-7] DNXHEGUUPJUMQT-UHFFFAOYSA-N | $2.6\times10^4$ | | HSDB (2015) | Q | 99 |
| estradiol $C_{18}H_{24}O_2$ [50-28-2] VOXZDWNPVJITMN-AWDGRILASA-N | $2.7\times10^5$ | | HSDB (2015) | Q | 99 |
| estriol $C_{18}H_{24}O_3$ [50-27-1] PROQIPRRNZUXQM-PVGHXWSTSA-N | $7.6\times10^6$ | | HSDB (2015) | Q | 99 |
| nandrolone $C_{18}H_{26}O_2$ [434-22-0] NPAGDVCDWIYMMC-SVXFNXITSA-N | $3.7\times10^3$ | | HSDB (2015) | Q | 99 |
| diofenolan $C_{18}H_{20}O_4$ [63837-33-2] ZDOOQPFIGYHZFV-UHFFFAOYSA-N | $1.5\times10^2$ $1.2\times10^3$ $1.5\times10^2$ | | Duchowicz et al. (2020) Duchowicz et al. (2020) MacBean (2012a) | V Q ? | 186 |
| dicumarol $C_{19}H_{12}O_6$ [66-76-2] DOBMPNYZJYQDGZ-UHFFFAOYSA-N | $7.0\times10^7$ | | HSDB (2015) | Q | 99 |
| coumatetralyl $C_{19}H_{16}O_3$ [5836-29-3] ULSLJYXHZDTLQK-UHFFFAOYSA-N | $1.7\times10^8$ | | HSDB (2015) | V | |
| warfarin $C_{19}H_{16}O_4$ [81-81-2] PJVWKTKQMONHTI-UHFFFAOYSA-N | $3.7\times10^4$ $3.6\times10^2$ | | HSDB (2015) Mackay et al. (2006d) | V V | |





Table A3.12: Miscellaneous (. . . continued)

| Substance Formula (Trivial Name) [CAS Registry Number] InChIKey | $H_s^{cp}$ (at $T^{\ominus}$) $\left[\dfrac{\mathrm{mol}}{\mathrm{m^3\,Pa}}\right]$ | $\dfrac{\mathrm{d}\ln H_s^{cp}}{\mathrm{d}(1/T)}$ [K] | Reference | Type | Note |
|---|---|---|---|---|---|
| tanshinone II $C_{19}H_{18}O_3$ [568-72-9] HYXITZLLTYIPOF-UHFFFAOYSA-N | $2.0\times10^3$ | | HSDB (2015) | Q | 99 |
| gibberellic acid $C_{19}H_{22}O_6$ [77-06-5] IXORZMNAPKEEDV-UHFFFAOYSA-N | $6.2\times10^9$ $1.3\times10^6$ | | HSDB (2015) Maniere et al. (2011) | Q ? | 99 165 |
| prallethrin $C_{19}H_{24}O_3$ [23031-36-9] SMKRKQBMYOFFMU-UHFFFAOYSA-N | 6.2 | | HSDB (2015) | V | |
| testolactone $C_{19}H_{24}O_3$ [968-93-4] BPEWUONYVDABNZ-LHXSAFEUSA-N | $1.6\times10^2$ | | HSDB (2015) | Q | 99 |
| androstenedione $C_{19}H_{26}O_2$ [63-05-8] AEMFNILZOJDQLW-JRCHKSGSSA-N | $2.7\times10^2$ | | HSDB (2015) | Q | 99 |
| diacetoxyscirpenol $C_{19}H_{26}O_7$ [2270-40-8] AUGQEEXBDZWUJY-UHFFFAOYSA-N | $1.0\times10^{11}$ | | HSDB (2015) | Q | 447 |
| testosterone $C_{19}H_{28}O_2$ [58-22-0] MUMGGOZAMZWBJJ-JZJKZLICSA-N | $2.8\times10^3$ | | HSDB (2015) | Q | 99 |
| $5\alpha$-androst-16-en-4-one $C_{19}H_{28}O$ (androstenone) [18339-16-7] HFVMLYAGWXSTQI-QYXZOKGRSA-N | $3.4\times10^{-2}$ | | Amoore and Buttery (1978) | M | |
| oxandrolone $C_{19}H_{30}O_3$ [53-39-4] QSLJIVKCVHQPLV-WPMSWULFSA-N | $4.3\times10^2$ | | HSDB (2015) | Q | 99 |
| piperonyl butoxide $C_{19}H_{30}O_5$ [51-03-6] FIPWRIJSWJWJAI-UHFFFAOYSA-N | $1.1\times10^5$ | | HSDB (2015) | Q | 99 |



Table A3.12: Miscellaneous (...continued)

| Substance Formula (Trivial Name) [CAS Registry Number] InChIKey | $H_s^{cp}$ (at $T^\ominus$) $\left[\dfrac{\mathrm{mol}}{\mathrm{m}^3\,\mathrm{Pa}}\right]$ | $\dfrac{\mathrm{d}\ln H_s^{cp}}{\mathrm{d}(1/T)}$ [K] | Reference | Type | Note |
|---|---|---|---|---|---|
| methoprene $C_{19}H_{34}O_3$ [40596-69-8] NFGXHKASABOEEW-LDRANXPESA-N | 1.4 | | HSDB (2015) | V | |
| fluorescein $C_{20}H_{12}O_5$ [2321-07-5] GNBHRKFJIUUOQI-UHFFFAOYSA-N | $1.1\times10^{11}$ | | HSDB (2015) | Q | 99 |
| phenolphthalein $C_{20}H_{14}O_4$ [77-09-8] KJFMBFZCATUALV-UHFFFAOYSA-N | $1.1\times10^{10}$ | | HSDB (2015) | Q | 99 |
| avobenzone $C_{20}H_{22}O_3$ [70356-09-1] XNEFYCZVKIDDMS-UHFFFAOYSA-N | $4.9\times10^{4}$ | | HSDB (2015) | Q | 447 |
| ethinyl estradiol $C_{20}H_{24}O_2$ [57-63-6] BFPYWIDHMRZLRN-UHFFFAOYSA-N | $1.2\times10^{6}$ | | HSDB (2015) | Q | 99 |
| norethynodrel $C_{20}H_{26}O_2$ [68-23-5] ICTXHFFSOAJUMG-CEVCPLMDSA-N | $7.6\times10^{3}$ | | HSDB (2015) | Q | 99 |
| norethindrone $C_{20}H_{26}O$ [68-22-4] VIKNJXKGJWUCNN-BROHZWGRSA-N | $1.7\times10^{4}$ | | HSDB (2015) | Q | 99 |
| methandrostenolone $C_{20}H_{28}O_2$ [72-63-9] XWALNWXLMVGSFR-NSDIEPNESA-N | $4.5\times10^{3}$ | | HSDB (2015) | Q | 99 |
| cinerin I $C_{20}H_{28}O_3$ [25402-06-6] FMTFEIJHMMQUJI-FPLPWBNLSA-N | $1.0\times10^{1}$ | | HSDB (2015) | Q | 99 |
| 17-methyltestosterone $C_{20}H_{30}O_2$ [58-18-4] GCKMFJBGXUYNAG-NSDIEPNESA-N | $2.1\times10^{3}$ | | HSDB (2015) | Q | 99 |





Table A3.12: Miscellaneous (... continued)

| Substance Formula (Trivial Name) [CAS Registry Number] InChIKey | $H_s^{cp}$ (at $T^{\ominus}$) $\left[\dfrac{\text{mol}}{\text{m}^3\,\text{Pa}}\right]$ | $\dfrac{\text{d}\ln H_s^{cp}}{\text{d}(1/T)}$ [K] | Reference | Type | Note |
|---|---|---|---|---|---|
| drostanolone C$_{20}$H$_{32}$O$_2$ (dromostanolone) [58-19-5] IKXILDNPCZPPRV-UHFFFAOYSA-N | $1.2\times10^3$ | | HSDB (2015) | Q | 99 |
| curcumin C$_{21}$H$_{20}$O$_6$ [458-37-7] VFLDPWHFBUODDF-FCXRPNKRSA-N | $1.4\times10^{16}$ | | HSDB (2015) | Q | 99 |
| bisphenol A diglycidyl ether C$_{21}$H$_{24}$O$_4$ [1675-54-3] LCFVJGUPQDGYKZ-UHFFFAOYSA-N | $2.2\times10^5$ | | HSDB (2015) | Q | 99 |
| mestranol C$_{21}$H$_{26}$O$_2$ [72-33-3] IMSSROKUHAOUJS-ALAWOQLPSA-N | $2.2\times10^3$ | | HSDB (2015) | Q | 99 |
| prednisone C$_{21}$H$_{26}$O$_5$ [53-03-2] XOFYZVNMUHMLCC-NUBBXXJJSA-N | $3.5\times10^4$ | | HSDB (2015) | Q | 99 |
| norgestrel C$_{21}$H$_{28}$O$_2$ [6533-00-2] WWYNJERNGUHSAO-ZUHHCLADSA-N | $1.3\times10^4$ | | HSDB (2015) | Q | 99 |
| levonorgestrel C$_{21}$H$_{28}$O$_2$ [797-63-7] WWYNJERNGUHSAO-XUDSTZEESA-N | $1.3\times10^4$ | | HSDB (2015) | Q | 99 |
| pyrethrin I C$_{21}$H$_{28}$O$_3$ [121-21-1] ROVGZAWFACYCSP-CMDGGOBGSA-N | $1.3\times10^1$ $2.2\times10^{-1}$ $8.4\times10^{-1}$ | | Duchowicz et al. (2020) HSDB (2015) Duchowicz et al. (2020) | V V Q | 186 |
| cinerin II C$_{21}$H$_{28}$O$_5$ [121-20-0] SHCRDCOTRILILT-WOBDGSLYSA-N | $1.1\times10^4$ | | HSDB (2015) | Q | 99 |
| prednisolone C$_{21}$H$_{28}$O$_5$ [50-24-8] OIGNJSKKLXVSLS-UHFFFAOYSA-N | $3.7\times10^2$ | | HSDB (2015) | Q | 99 |





Table A3.12: Miscellaneous (... continued)

| Substance Formula (Trivial Name) [CAS Registry Number] InChIKey | $H_s^{cp}$ (at $T^\ominus$) $\left[\dfrac{\mathrm{mol}}{\mathrm{m^3\,Pa}}\right]$ | $\dfrac{\mathrm{d}\ln H_s^{cp}}{\mathrm{d}(1/T)}$ [K] | Reference | Type | Note |
|---|---|---|---|---|---|
| dronabinol $C_{21}H_{30}O_2$ (delta 9-tetrahydrocannabinol) [1972-08-3] CYQFCXCEBYINGO-UHFFFAOYSA-N | $4.1\times10^1$ | | HSDB (2015) | Q | 99 |
| progesterone $C_{21}H_{30}O_2$ [57-83-0] RJKFOVLPORLFTN-UHFFFAOYSA-N | $1.5\times10^2$ | | HSDB (2015) | Q | 99 |
| hydrocortisone $C_{21}H_{30}O_5$ [50-23-7] JYGXADMDTFJGBT-NDNUHCHRSA-N | $1.7\times10^2$ | | HSDB (2015) | Q | 99 |
| calusterone $C_{21}H_{32}O_2$ [17021-26-0] IVFYLRMMHVYGJH-UHFFFAOYSA-N | $1.6\times10^3$ | | HSDB (2015) | Q | 99 |
| oxymetholone $C_{21}H_{32}O_3$ [434-07-1] ICMWWNHDUZJFDW-RCXBLOTCSA-N | $6.6\times10^3$ | | HSDB (2015) | Q | 99 |
| resmethrin $C_{22}H_{26}O_3$ [10453-86-8] VEMKTZHHVJILDY-UHFFFAOYSA-N | $4.7\times10^{-2}$ $7.6\times10^1$ $2.2\times10^{-1}$ | | Duchowicz et al. (2020) HSDB (2015) Duchowicz et al. (2020) | V V Q | 186 |
| pyrethrin II $C_{22}H_{28}O_5$ [121-29-9] VJFUPGQZSXIULQ-VKTMSVCMSA-N | $4.5\times10^2$ $4.5\times10^2$ $3.8\times10^2$ | | Duchowicz et al. (2020) HSDB (2015) Duchowicz et al. (2020) | V V Q | 186 |
| methylprednisolone $C_{22}H_{30}O_5$ [83-43-2] VHRSUDSXCMQTMA-UHFFFAOYSA-N | $2.7\times10^2$ | | HSDB (2015) | Q | 99 |
| medroxyprogesterone $C_{22}H_{32}O_3$ [520-85-4] FRQMUZJSZHZSGN-HBNHAYAOSA-N | $7.6\times10^2$ | | HSDB (2015) | Q | 99 |
| dimethirimol $C_{23}H_{24}O_5$ [5221-53-4] VNKCZJKGJAEOCW-WXUKJITCSA-N | $>2.3\times10^{10}$ | | MacBean (2012a) | ? | |





Table A3.12: Miscellaneous (...continued)

| Substance Formula (Trivial Name) [CAS Registry Number] InChIKey | $H_s^{cp}$ (at $T^{\ominus}$) $\left[\dfrac{\mathrm{mol}}{\mathrm{m^3\,Pa}}\right]$ | $\dfrac{\mathrm{d}\ln H_s^{cp}}{\mathrm{d}(1/T)}$ [K] | Reference | Type | Note |
|---|---|---|---|---|---|
| rotenone $C_{23}H_{22}O_6$ [83-79-4] JUVIOZPCNVVQFO-UHFFFAOYSA-N | $8.8\times10^7$ | | HSDB (2015) | Q | 99 |
| phenothrin $C_{23}H_{26}O_3$ [26002-80-2] SBNFWQZLDJGRLK-UHFFFAOYSA-N | 1.5 | | MacBean (2012b) | X | 350 |
| spiromesifen $C_{23}H_{30}O_4$ [283594-90-1] GOLXNESZZPUPJE-UHFFFAOYSA-N | $1.8\times10^{-2}$ | | HSDB (2015) | V | |
| digoxigenin $C_{23}H_{34}O_5$ [1672-46-4] SHIBSTMRCDJXLN-KCZCNTNESA-N | $4.3\times10^5$ | | HSDB (2015) | Q | 99 |
| annatto $C_{24}H_{28}O_4$ [1393-63-1] ZVKOASAVGLETCT-LRRSNBNMSA-N | $1.5\times10^{11}$ | | HSDB (2015) | Q | 99 |
| acequinocyl $C_{24}H_{32}O_4$ [57960-19-7] QDRXWCAVUNHOGA-UHFFFAOYSA-N | $1.0\times10^1$ $1.0\times10^1$ | | HSDB (2015) Maniere et al. (2011) | V ? | 241, 165 |
| T-2 mycotoxin $C_{24}H_{34}O_9$ [21259-20-1] SSHHYBPAMCLKRH-UHFFFAOYSA-N | $1.8\times10^{12}$ | | HSDB (2015) | Q | 99 |
| milk thistle extract $C_{25}H_{22}O_{10}$ [84604-20-6] SEBFKMXJBCUCAI-VGHNRKBZSA-N | $6.2\times10^{17}$ | | HSDB (2015) | Q | 99 |
| simvastatin $C_{25}H_{38}O_5$ [79902-63-9] RYMZZMVNJRMUDD-UHFFFAOYSA-N | $3.5\times10^4$ | | HSDB (2015) | Q | 99 |
| calcitriol $C_{27}H_{44}O_3$ (1,25-dihydroxycholecalciferol) [32222-06-3] GMRQFYUYWCNGIN-NKMMMXOESA-N | $3.2\times10^1$ | | HSDB (2015) | Q | 99 |



Table A3.12: Miscellaneous (...continued)

| Substance Formula (Trivial Name) [CAS Registry Number] InChIKey | $H_s^{cp}$ (at $T^\ominus$) $\left[\dfrac{\text{mol}}{\text{m}^3\,\text{Pa}}\right]$ | $\dfrac{\text{d}\ln H_s^{cp}}{\text{d}(1/T)}$ [K] | Reference | Type | Note |
|---|---|---|---|---|---|
| paricalcitol $C_{27}H_{44}O_3$ [131918-61-1] BPKAHTKRCLCHEA-UBFJEZKGSA-N | $2.6\times10^1$ | | HSDB (2015) | Q | 99 |
| cholecalciferol $C_{27}H_{44}O$ [67-97-0] QYSXJUFSXHHAJI-SMGPGMQOSA-N | $4.3\times10^{-2}$ | | HSDB (2015) | Q | 99 |
| cholesterol $C_{27}H_{46}O$ [57-88-5] HVYWMOMLDIMFJA-VUDDDUNTSA-N | $5.8\times10^{-2}$ | | HSDB (2015) | Q | 99 |
| ergosterol $C_{28}H_{44}O$ [57-87-4] DNVPQKQSNYMLRS-CVGROQQCSA-N | $6.2\times10^{-2}$ | | HSDB (2015) | Q | 99 |
| dihydrotachysterol $C_{28}H_{46}O$ [67-96-9] DTSXXSAWQHPLEF-GFVAUXBKSA-N | $2.7\times10^{-2}$ | | HSDB (2015) | Q | 99 |
| etoposide $C_{29}H_{32}O_{13}$ [33419-42-0] VJJPUSNTGOMMGY-KWGSHVRASA-N | $5.8\times10^{24}$ | | HSDB (2015) | Q | 99 |
| stigmasterol $C_{29}H_{48}O$ [83-48-7] HCXVJBMSMIARIN-BASBAMEESA-N | $3.8\times10^{-2}$ | | HSDB (2015) | Q | 99 |
| pseudohypericin $C_{30}H_{16}O_9$ [55954-61-5] YXBUQQDFTYOHQI-UHFFFAOYSA-N | $5.5\times10^{23}$ | | HSDB (2015) | Q | 99 |
| gossypol $C_{30}H_{30}O_8$ [303-45-7] QBKSWRVVCFFDOT-UHFFFAOYSA-N | $4.3\times10^{22}$ | | HSDB (2015) | Q | 99 |
| maslinic acid $C_{30}H_{48}O_4$ [4373-41-5] MDZKJHQSJHYOHJ-LLICELPBSA-N | $2.8\times10^5$ | | HSDB (2015) | Q | 447 |





Table A3.12: Miscellaneous (...continued)

| Substance Formula (Trivial Name) [CAS Registry Number] InChIKey | $H_s^{cp}$ (at $T^\ominus$) $\left[\dfrac{\text{mol}}{\text{m}^3\,\text{Pa}}\right]$ | $\dfrac{\text{d}\ln H_s^{cp}}{\text{d}(1/T)}$ [K] | Reference | Type | Note |
|---|---|---|---|---|---|
| milbemectin $C_{31}H_{44}O_7$ [51596-10-2] ZLBGSRMUSVULIE-GSMJGMFJSA-N | $6.5\times10^2$ $3.9\times10^2$ | | Maniere et al. (2011) Maniere et al. (2011) | ? ? | 241, 165 241, 165 |
| difenacoum $C_{31}H_{24}O_3$ [56073-07-5] FVQITOLOYMWVFU-UHFFFAOYSA-N | $7.0\times10^6$ | | HSDB (2015) | Q | 447 |
| nonoxynol 9 $C_{33}H_{60}O_{10}$ [26571-11-9] FBWNMEQMRUMQSO-UHFFFAOYSA-N | $1.8\times10^{16}$ | | HSDB (2015) | Q | 99 |
| azadirachtin $C_{35}H_{44}O_{16}$ [11141-17-6] FTNJWQUOZFUQQJ-GWTPYEAISA-N | $3.5\times10^{19}$ | | HSDB (2015) | V | |
| monensin $C_{36}H_{62}O_{11}$ [17090-79-8] GAOZTHIDHYLHMS-LXKLZWMJSA-N | $4.9\times10^{18}$ | | HSDB (2015) | Q | 99 |
| gossyplure $C_{36}H_{64}O_4$ [50933-33-0] BXJHOKLLMOYSRQ-QOXWLJPHSA-N | $6.6\times10^{-2}$ | | HSDB (2015) | V | |
| capsanthin $C_{40}H_{56}O_3$ [465-42-9] VYIRVAXUEZSDNC-RDJLEWNRSA-N | $3.4\times10^2$ | | HSDB (2015) | Q | 99 |
| heptamaloxyloglucan $C_{40}H_{70}O_{33}$ [870721-81-6] RAUODYOTTYNEJP-RQESCVSBSA-N | $4.2\times10^{13}$ | | Maniere et al. (2011) | ? | 12, 165 |
| digitoxin $C_{41}H_{64}O_{13}$ [71-63-6] WDJUZGPOPHTGOT-UCKSZOHFSA-N | $7.6\times10^{19}$ | | HSDB (2015) | Q | 99 |
| digoxin $C_{41}H_{64}O_{14}$ [20830-75-5] LTMHDMANZUZIPE-PUGKRICDSA-N | $2.1\times10^{21}$ | | HSDB (2015) | Q | 99 |



Table A3.12: Miscellaneous (...continued)

| Substance<br>Formula<br>(Trivial Name)<br>[CAS Registry Number]<br>InChIKey | $H_s^{cp}$<br>(at $T^{\ominus}$)<br>$\left[\dfrac{\text{mol}}{\text{m}^3\,\text{Pa}}\right]$ | $\dfrac{\text{d}\ln H_s^{cp}}{\text{d}(1/T)}$<br><br>[K] | Reference | Type | Note |
|---|---|---|---|---|---|
| pyrethrum<br>$C_{43}H_{56}O_8$<br>[8003-34-7]<br>VXSIXFKKSNGRRO-YWUDCVDHSA-N | $1.5\times10^1$<br>$1.4\times10^4$<br>$1.3\times10^1$ | | HSDB (2015)<br>Maniere et al. (2011)<br>Maniere et al. (2011) | Q<br>?<br>? | 99<br>165<br>165 |
| punicalagin<br>$C_{48}H_{28}O_{30}$<br>[65995-63-3]<br>SKNLUADAGHCXKF-UHFFFAOYSA-N | $5.5\times10^{10}$ | | HSDB (2015) | Q | 99 |
| abamectin<br>$C_{48}H_{72}O_{14}$<br>[71751-41-2]<br>RRZXIRBKKLTSOM-IGNCFFBFSA-N | $7.0\times10^3$<br>$>3.7\times10^2$ | | HSDB (2015)<br>Maniere et al. (2011) | V<br>? | <br>12, 165 |
| notoginsenoside R1<br>$C_{48}H_{84}O_{18}$<br>[80418-24-2]<br>ZBXDHDDTAIOMHK-UWGJVBKGSA-N | $6.6\times10^{25}$ | | HSDB (2015) | Q | 99 |
| triolein<br>$C_{57}H_{104}O_6$<br>[122-32-7]<br>PHYFQTYBJUILEZ-IUPFWZBJSA-N | $1.0\times10^{-2}$ | | HSDB (2015) | Q | 447 |
| tristearin<br>$C_{57}H_{110}O_6$<br>[555-43-1]<br>DCXXMTOCNZCJGO-UHFFFAOYSA-N | $7.0\times10^{-3}$ | | HSDB (2015) | Q | 99 |



## A4 Organic species with nitrogen (N)

### A4.1 Amines (C, H, N)

Table A4.1: Amines (C, H, N)

| Substance Formula (Trivial Name) [CAS Registry Number] InChIKey | $H_s^{cp}$ (at $T^{\ominus}$) $\left[\dfrac{\mathrm{mol}}{\mathrm{m^3\,Pa}}\right]$ | $\dfrac{\mathrm{d}\ln H_s^{cp}}{\mathrm{d}(1/T)}$ [K] | Reference | Type | Note |
|---|---|---|---|---|---|
| cyanamide $CH_2N_2$ [420-04-2] XZMCDFZZKTWFGF-UHFFFAOYSA-N | $3.7\times10^4$ | | HSDB (2015) | V | |
| methylhydrazine $CH_6N_2$ [60-34-4] HDZGCSFEDULWCS-UHFFFAOYSA-N | 3.3 | | HSDB (2015) | V | |
| methanamine $CH_3NH_2$ (methylamine) [74-89-5] BAVYZALUXZFZLV-UHFFFAOYSA-N | $8.9\times10^{-1}$ | | Burkholder et al. (2019) | L | |
| | $8.9\times10^{-1}$ | | Burkholder et al. (2015) | L | |
| | $3.5\times10^{-1}$ | 2600 | Wilhelm et al. (1977) | L | |
| | $8.9\times10^{-1}$ | | Christie and Crisp (1967) | M | |
| | 1.1 | | Yaws (2003) | X | 237 |
| | $7.4\times10^{-1}$ | | Keshavarz et al. (2022) | Q | |
| | 6.2 | | Duchowicz et al. (2020) | Q | |
| | 1.1 | | Gharagheizi et al. (2010) | Q | 246 |
| | 1.2 | | Hilal et al. (2008) | Q | |
| | 2.3 | | Modarresi et al. (2007) | Q | 67 |
| | | 5000 | Kühne et al. (2005) | Q | |
| | $9.2\times10^{-1}$ | | Yaffe et al. (2003) | Q | 248, 249 |
| | 1.1 | | English and Carroll (2001) | Q | 230, 231 |
| | $6.1\times10^{-1}$ | | Katritzky et al. (1998) | Q | |
| | $5.6\times10^{-1}$ | | Nirmalakhandan et al. (1997) | Q | |
| | $8.9\times10^{-1}$ | | Duchowicz et al. (2020) | ? | 185, 21 |
| | $8.9\times10^{-1}$ | | Mackay et al. (2006d) | ? | |
| | | 3200 | Kühne et al. (2005) | ? | |
| | $8.8\times10^{-1}$ | | Abraham et al. (1990) | ? | |
| | | 5400 | Abraham (1984) | ? | 21 |
| | 1.4 | | Bone et al. (1983) | ? | 65 |
| ethanamine $C_2H_5NH_2$ (ethylamine) [75-04-7] QUSNBJAOOMFDIB-UHFFFAOYSA-N | $8.0\times10^{-1}$ | | Burkholder et al. (2019) | L | |
| | $8.0\times10^{-1}$ | | Burkholder et al. (2015) | L | |
| | $8.4\times10^{-1}$ | | Brockbank (2013) | L | |
| | $3.5\times10^{-1}$ | 3600 | Wilhelm et al. (1977) | L | |
| | $8.0\times10^{-1}$ | | Christie and Crisp (1967) | M | |
| | $9.9\times10^{-1}$ | | Butler and Ramchandani (1935) | M | |
| | $3.0\times10^{-1}$ | | Hwang et al. (1992) | V | |
| | 1.0 | | Keshavarz et al. (2022) | Q | |
| | 2.2 | | Duchowicz et al. (2020) | Q | 299 |
| | $9.7\times10^{-1}$ | | Li et al. (2014) | Q | 241 |
| | $7.9\times10^{-1}$ | | Hilal et al. (2008) | Q | |
| | 1.7 | | Modarresi et al. (2007) | Q | 67 |




Table A4.1: Amines (C, H, N) (... continued)

| Substance<br>Formula<br>(Trivial Name)<br>[CAS Registry Number]<br>InChIKey | $H_s^{cp}$<br>(at $T^{\ominus}$)<br>$\left[\dfrac{\mathrm{mol}}{\mathrm{m^3\,Pa}}\right]$ | $\dfrac{\mathrm{d}\ln H_s^{cp}}{\mathrm{d}(1/T)}$<br><br>[K] | Reference | Type | Note |
|---|---|---|---|---|---|
| | $3.3\times10^{-1}$ | | Yao et al. (2002) | Q | 229 |
| | $9.5\times10^{-1}$ | | English and Carroll (2001) | Q | 230, 231 |
| | $4.7\times10^{-1}$ | | Katritzky et al. (1998) | Q | |
| | $4.6\times10^{-1}$ | | Nirmalakhandan et al. (1997) | Q | |
| | 2.4 | | Russell et al. (1992) | Q | 279 |
| | 1.3 | | Suzuki et al. (1992) | Q | 232 |
| | $8.0\times10^{-1}$ | | Duchowicz et al. (2020) | ? | 185, 21 |
| | $9.9\times10^{-1}$ | | Mackay et al. (2006d) | ? | |
| | $3.8\times10^{-1}$ | | Yaws (1999) | ? | 21, 12 |
| | $8.0\times10^{-1}$ | | Abraham et al. (1990) | ? | |
| | | 6500 | Abraham (1984) | ? | 21 |
| 1H-1,2,4-triazole<br>$C_2H_3N_3$<br>[288-88-0]<br>NSPMIYGKQJPBQR-UHFFFAOYSA-N | 6.6 | | HSDB (2015) | Q | 99 |
| dicyandiamide<br>$C_2H_4N_4$<br>(cyanoguanidine)<br>[461-58-5]<br>QGBSISYHAICWAH-UHFFFAOYSA-N | $4.3\times10^4$ | | HSDB (2015) | Q | 99 |
| ethylenimine<br>$C_2H_5N$<br>[151-56-4]<br>NOWKCMXCCJGMRR-UHFFFAOYSA-N | $8.2\times10^{-1}$<br>$8.2\times10^{-1}$<br>$2.9\times10^1$ | | Duchowicz et al. (2020)<br>HSDB (2015)<br>Duchowicz et al. (2020) | V<br>V<br>Q | 186 |
| 1,2-dimethylhydrazine<br>$C_2H_8N_2$<br>[540-73-8]<br>DIIIISSCIXVANO-UHFFFAOYSA-N | 1.8 | | HSDB (2015) | V | |
| 1,1-dimethylhydrazine<br>$C_2H_8N_2$<br>[57-14-7]<br>RHUYHJGZWVXEHW-UHFFFAOYSA-N | $7.6\times10^{-1}$ | | HSDB (2015) | V | |
| 1-propanamine<br>$C_3H_7NH_2$<br>(propylamine)<br>[107-10-8]<br>WGYKZJWCGVVSQN-UHFFFAOYSA-N | $6.6\times10^{-1}$<br>$6.6\times10^{-1}$<br>$6.7\times10^{-1}$<br>$5.6\times10^{-1}$<br>$5.0\times10^{-1}$<br>$6.6\times10^{-1}$<br>$7.8\times10^{-1}$<br>$6.6\times10^{-1}$<br>1.4<br>2.3<br>$4.8\times10^{-1}$<br>2.1 | <br><br><br>6400<br><br><br><br>6700 | Burkholder et al. (2019)<br>Burkholder et al. (2015)<br>Brockbank (2013)<br>Leng et al. (2015a)<br>Altschuh et al. (1999)<br>Christie and Crisp (1967)<br>Butler and Ramchandani (1935)<br>Plyasunov et al. (2001)<br>Keshavarz et al. (2022)<br>Duchowicz et al. (2020)<br>Hilal et al. (2008)<br>Modarresi et al. (2007) | L<br>L<br>L<br>M<br>M<br>M<br>M<br>T<br>Q<br>Q<br>Q<br>Q | <br><br><br><br><br><br><br><br><br>184<br><br>67 |



Table A4.1: Amines (C, H, N) (. . . continued)

| Substance<br>Formula<br>(Trivial Name)<br>[CAS Registry Number]<br>InChIKey | $H_s^{cp}$<br>(at $T^{\ominus}$)<br>$\left[\dfrac{\text{mol}}{\text{m}^3\,\text{Pa}}\right]$ | $\dfrac{\text{d}\ln H_s^{cp}}{\text{d}(1/T)}$<br><br>[K] | Reference | Type | Note |
|---|---|---|---|---|---|
| | $6.7\times10^{-1}$ | | Yaffe et al. (2003) | Q | 248, 249 |
| | $2.6\times10^{-1}$ | | Yao et al. (2002) | Q | 229 |
| | $5.2\times10^{-1}$ | | Katritzky et al. (1998) | Q | |
| | $3.6\times10^{-1}$ | | Nirmalakhandan et al. (1997) | Q | |
| | 1.6 | | Russell et al. (1992) | Q | 279 |
| | 1.0 | | Suzuki et al. (1992) | Q | 232 |
| | $6.7\times10^{-1}$ | | Duchowicz et al. (2020) | ? | 185, 21 |
| | $7.8\times10^{-1}$ | | Mackay et al. (2006d) | ? | |
| | $3.9\times10^{-1}$ | | Yaws (1999) | ? | 21 |
| | $6.7\times10^{-1}$ | | Abraham et al. (1990) | ? | |
| | | 6700 | Abraham (1984) | ? | 21 |
| 2-propanamine<br>$C_3H_9N$<br>(isopropylamine)<br>[75-31-0]<br>JJWLVOIRVHMVIS-UHFFFAOYSA-N | $2.2\times10^{-1}$ | | Duchowicz et al. (2020) | V | 186 |
| | $2.2\times10^{-1}$ | | Hilal et al. (2008) | C | |
| | $8.9\times10^{-1}$ | | Duchowicz et al. (2020) | Q | |
| | $2.1\times10^{-1}$ | | Hilal et al. (2008) | Q | |
| | 2.4 | | Modarresi et al. (2007) | Q | 67 |
| | $2.3\times10^{-1}$ | | Yaffe et al. (2003) | Q | 248, 249 |
| | $1.6\times10^{-1}$ | | Yao et al. (2002) | Q | 229 |
| | $3.1\times10^{-1}$ | | Katritzky et al. (1998) | Q | |
| | $1.3\times10^{-1}$ | | Yaws (1999) | ? | 21, 12 |
| propanedinitrile<br>$C_3H_2N_2$<br>(malononitrile)<br>[109-77-3]<br>CUONGYYJJVDODC-UHFFFAOYSA-N | $7.5\times10^{1}$ | | Duchowicz et al. (2020) | V | 186 |
| | $2.0\times10^{1}$ | | Duchowicz et al. (2020) | Q | |
| | $7.8\times10^{2}$ | | HSDB (2015) | Q | 99 |
| | $2.0\times10^{1}$ | | Gharagheizi et al. (2012) | Q | |
| | $6.3\times10^{1}$ | | Yaws (1999) | ? | 21, 12 |
| 2-methylaziridine<br>$C_3H_7N$<br>[75-55-8]<br>OZDGMOYKSFPLSE-UHFFFAOYSA-N | $9.9\times10^{-1}$ | | Duchowicz et al. (2020) | V | 186 |
| | $1.2\times10^{1}$ | | Duchowicz et al. (2020) | Q | |
| | 1.2 | | Yaffe et al. (2003) | Q | 248, 249 |
| | 1.5 | | Katritzky et al. (1998) | Q | |
| 1,2-diaminopropane<br>$C_3H_{10}N_2$<br>[78-90-0]<br>AOHJOMMDDJHIJH-UHFFFAOYSA-N | $1.4\times10^{3}$ | 7400 | Nguyen (2013) | M | 11 |
| 1-butanamine<br>$C_4H_9NH_2$<br>(butylamine)<br>[109-73-9]<br>HQABUPZFAYXKJW-UHFFFAOYSA-N | $5.6\times10^{-1}$ | | Burkholder et al. (2019) | L | |
| | $5.6\times10^{-1}$ | | Burkholder et al. (2015) | L | |
| | $5.4\times10^{-1}$ | | Brockbank (2013) | L | |
| | $5.6\times10^{-1}$ | | Altschuh et al. (1999) | M | |
| | $5.2\times10^{-1}$ | | Rytting et al. (1978) | M | |
| | $5.6\times10^{-1}$ | | Christie and Crisp (1967) | M | |
| | $6.5\times10^{-1}$ | | Butler and Ramchandani (1935) | M | |
| | $2.2\times10^{-1}$ | | Hwang et al. (1992) | V | |
| | $4.5\times10^{-1}$ | | Amoore and Buttery (1978) | V | |
| | $3.7\times10^{-1}$ | | Yaws (2003) | X | 258 |
| | $3.6\times10^{-1}$ | | Yaws (2003) | X | 237 |
| | $2.1\times10^{-1}$ | | Dupeux et al. (2022) | Q | 259 |



Table A4.1: Amines (C, H, N) (. . . continued)

| Substance Formula (Trivial Name) [CAS Registry Number] InChIKey | $H_s^{cp}$ (at $T^{\ominus}$) $\left[\dfrac{\text{mol}}{\text{m}^3\,\text{Pa}}\right]$ | $\dfrac{\text{d}\ln H_s^{cp}}{\text{d}(1/T)}$ [K] | Reference | Type | Note |
|---|---|---|---|---|---|
|  | $1.2\times10^{-1}$ |  | Keshavarz et al. (2022) | Q |  |
|  | $2.3$ |  | Duchowicz et al. (2020) | Q | 299 |
|  | $6.5\times10^{-1}$ |  | Li et al. (2014) | Q | 241 |
|  | $4.0\times10^{-1}$ |  | Gharagheizi et al. (2010) | Q | 246 |
|  | $2.9\times10^{-1}$ |  | Hilal et al. (2008) | Q |  |
|  | $1.5$ |  | Modarresi et al. (2007) | Q | 67 |
|  | $5.2\times10^{-1}$ |  | Yaffe et al. (2003) | Q | 248, 249 |
|  | $2.4\times10^{-1}$ |  | Yao et al. (2002) | Q | 229 |
|  | $7.3\times10^{-1}$ |  | English and Carroll (2001) | Q | 230, 231 |
|  | $4.0\times10^{-1}$ |  | Katritzky et al. (1998) | Q |  |
|  | $2.8\times10^{-1}$ |  | Nirmalakhandan et al. (1997) | Q |  |
|  | $1.1$ |  | Russell et al. (1992) | Q | 279 |
|  | $7.9\times10^{-1}$ |  | Suzuki et al. (1992) | Q | 232 |
|  | $5.7\times10^{-1}$ |  | Duchowicz et al. (2020) | ? | 185, 21 |
|  | $6.6\times10^{-1}$ |  | Mackay et al. (2006d) | ? |  |
|  | $2.2\times10^{-1}$ |  | Yaws (1999) | ? | 21 |
|  | $5.2\times10^{-1}$ |  | Abraham et al. (1990) | ? |  |
|  |  | 7100 | Abraham (1984) | ? | 21 |
| 2-butanamine C$_4$H$_{11}$N (*sec*-butylamine) [13952-84-6] BHRZNVHARXXAHW-UHFFFAOYSA-N | $4.0\times10^{-1}$ | 7700 | Kish et al. (2013) | M | 548 |
|  | $6.5\times10^{-2}$ |  | Duchowicz et al. (2020) | V | 186 |
|  | $6.5\times10^{-2}$ |  | Hilal et al. (2008) | C |  |
|  | $9.1\times10^{-1}$ |  | Duchowicz et al. (2020) | Q |  |
|  | $1.6\times10^{-1}$ |  | Hilal et al. (2008) | Q |  |
|  | $1.7$ |  | Modarresi et al. (2007) | Q | 67 |
|  | $2.2\times10^{-1}$ |  | Katritzky et al. (1998) | Q |  |
| 2-methyl-1-propanamine C$_4$H$_{11}$N (isobutylamine) [78-81-9] KDSNLYIMUZNERS-UHFFFAOYSA-N | $7.3\times10^{-1}$ |  | Duchowicz et al. (2020) | V | 186 |
|  | $7.2\times10^{-1}$ |  | Hilal et al. (2008) | C |  |
|  | $9.1\times10^{-1}$ |  | Duchowicz et al. (2020) | Q |  |
|  | $2.4\times10^{-1}$ |  | Hilal et al. (2008) | Q |  |
|  | $2.0$ |  | Modarresi et al. (2007) | Q | 67 |
|  | $5.2\times10^{-1}$ |  | Yaffe et al. (2003) | Q | 248, 272 |
|  | $1.5\times10^{-1}$ |  | Yao et al. (2002) | Q | 229, 267 |
|  | $3.9\times10^{-1}$ |  | Katritzky et al. (1998) | Q |  |
|  | $4.4\times10^{-1}$ |  | Yaws (1999) | ? | 21, 12 |
| 2-methyl-2-propanamine C$_4$H$_{11}$N (*tert*-butylamine) [75-64-9] YBRBMKDOPFTVDT-UHFFFAOYSA-N | $2.8\times10^{-1}$ |  | Duchowicz et al. (2020) | V | 186 |
|  | $2.8\times10^{-1}$ |  | Hilal et al. (2008) | C |  |
|  | $4.0\times10^{-1}$ |  | Duchowicz et al. (2020) | Q |  |
|  | $5.0\times10^{-2}$ |  | Hilal et al. (2008) | Q |  |
|  | $1.0\times10^{-1}$ |  | Modarresi et al. (2007) | Q | 67 |
|  | $2.9\times10^{-1}$ |  | Yaffe et al. (2003) | Q | 248, 249 |
|  | $2.1\times10^{-1}$ |  | Katritzky et al. (1998) | Q |  |
| dimethylethylamine C$_4$H$_{11}$N [598-56-1] DAZXVJBJRMWXJP-UHFFFAOYSA-N | $1.1\times10^{-1}$ |  | Yaws (2003) | X | 237 |
|  | $9.0\times10^{-2}$ |  | Gharagheizi et al. (2012) | Q |  |
|  | $1.1\times10^{-1}$ |  | Gharagheizi et al. (2010) | Q | 246 |



Table A4.1: Amines (C, H, N) (...continued)

| Substance Formula (Trivial Name) [CAS Registry Number] InChIKey | $H_s^{cp}$ (at $T^{\ominus}$) $\left[\dfrac{\mathrm{mol}}{\mathrm{m^3\,Pa}}\right]$ | $\dfrac{\mathrm{d}\ln H_s^{cp}}{\mathrm{d}(1/T)}$ [K] | Reference | Type | Note |
|---|---|---|---|---|---|
| 2-propylguanidine C$_4$H$_{11}$N$_3$ [462-25-9] BWMDMTSNSXYYSP-UHFFFAOYSA-N | $4.1\times10^4$ | | Ebert et al. (2023) | ? | 365 |
| 1,2-diethylhydrazine C$_4$H$_{12}$N$_2$ [1615-80-1] YCBOYOYVDOUXLH-UHFFFAOYSA-N | $8.2\times10^1$ | | HSDB (2015) | Q | 99 |
| N-(2-aminoethyl)-1,2-ethanediamine C$_4$H$_{13}$N$_3$ (diethylenetriamine) [111-40-0] RPNUMPOLZDHAAY-UHFFFAOYSA-N | $9.9\times10^8$ | | HSDB (2015) | Q | 99 |
| butanedinitrile C$_4$H$_4$N$_2$ [110-61-2] IAHFWCOBPZCAEA-UHFFFAOYSA-N | $8.7\times10^2$ $1.5\times10^3$ $1.5\times10^3$ $4.0\times10^3$ $2.6\times10^1$ $3.9\times10^3$ | 7000 | Plyasunov et al. (2006) Duchowicz et al. (2020) HSDB (2015) Yaws (2003) Duchowicz et al. (2020) Gharagheizi et al. (2010) | L V V X Q Q | 186 237, 12 246 |
| 2-methyl-1H-imidazole C$_4$H$_6$N$_2$ [693-98-1] LXBGSDVWAMZHDD-UHFFFAOYSA-N | $8.1\times10^2$ $1.8\times10^3$ $1.1\times10^3$ $2.2$ | | Du et al. (2017) Du et al. (2017) Du et al. (2017) HSDB (2015) | M Q Q Q | 478 549 99 |
| 4-methyl-1H-imidazole C$_4$H$_6$N$_2$ [822-36-6] XLSZMDLNRCVEIJ-UHFFFAOYSA-N | $2.4$ | | HSDB (2015) | Q | 99 |
| N-methyl-1,3-propanediamine C$_4$H$_{12}$N$_2$ (3-(methylamino)propylamine) [6291-84-5] QHJABUZHRJTCAR-UHFFFAOYSA-N | $1.4\times10^3$ $2.1\times10^3$ | 7600 8800 | Nguyen (2013) Kim et al. (2008) | M M | 11 550 |
| 1-pentanamine C$_5$H$_{11}$NH$_2$ (1-pentylamine) [110-58-7] DPBLXKKOBLCELK-UHFFFAOYSA-N | $4.3\times10^{-1}$ $4.0\times10^{-1}$ $3.1\times10^{-1}$ $4.0\times10^{-1}$ $1.7\times10^{-1}$ $2.4$ $1.6\times10^{-1}$ $1.2$ $4.0\times10^{-1}$ $5.6\times10^{-1}$ | | Brockbank (2013) Rytting et al. (1978) Amoore and Buttery (1978) Christie and Crisp (1967) Keshavarz et al. (2022) Duchowicz et al. (2020) Hilal et al. (2008) Modarresi et al. (2007) Yaffe et al. (2003) English and Carroll (2001) | L M M M Q Q Q Q Q Q | 299 67 248, 249 230, 260 |





Table A4.1: Amines (C, H, N) (... continued)

| Substance<br>Formula<br>(Trivial Name)<br>[CAS Registry Number]<br>InChIKey | $H_s^{cp}$<br>(at $T^{\ominus}$)<br>$\left[\dfrac{\mathrm{mol}}{\mathrm{m^3\,Pa}}\right]$ | $\dfrac{\mathrm{d}\ln H_s^{cp}}{\mathrm{d}(1/T)}$<br><br>[K] | Reference | Type | Note |
|---|---|---|---|---|---|
| | $4.0\times10^{-1}$ | | Katritzky et al. (1998) | Q | |
| | $2.2\times10^{-1}$ | | Nirmalakhandan et al. (1997) | Q | |
| | $7.2\times10^{-1}$ | | Russell et al. (1992) | Q | 358 |
| | $1.5\times10^{-1}$ | | Suzuki et al. (1992) | Q | 232 |
| | $4.1\times10^{-1}$ | | Duchowicz et al. (2020) | ? | 185, 21 |
| | $4.0\times10^{-1}$ | | Abraham et al. (1990) | ? | |
| | | 7500 | Abraham (1984) | ? | 21 |
| 3-methyl-1-butanamine<br>$C_5H_{13}N$<br>[107-85-7]<br>BMFVGAAISNGQNM-UHFFFAOYSA-N | $3.2\times10^{-1}$<br>$3.1\times10^{-1}$<br>$2.2\times10^{-1}$ | | Yaws (2003)<br>Gharagheizi et al. (2010)<br>Hilal et al. (2008) | X<br>Q<br>Q | 237<br>246 |
| 1,2-dimethylpropylamine<br>$C_5H_{13}N$<br>[598-74-3]<br>JOZZAIIGWFLONA-UHFFFAOYSA-N | $4.0\times10^{-1}$<br>$3.8\times10^{-1}$ | | Yaws (2003)<br>Gharagheizi et al. (2010) | X<br>Q | 237<br>246 |
| dimethylpropylamine<br>$C_5H_{13}N$<br>[926-63-6]<br>ZUHZZVMEUAUWHY-UHFFFAOYSA-N | $7.9\times10^{-2}$<br>$5.3\times10^{-2}$<br>$9.0\times10^{-2}$ | | Yaws (2003)<br>Gharagheizi et al. (2012)<br>Gharagheizi et al. (2010) | X<br>Q<br>Q | 237<br><br>246 |
| dimethylisopropylamine<br>$C_5H_{13}N$<br>[996-35-0]<br>VMOWKUTXPNPTEN-UHFFFAOYSA-N | $7.9\times10^{-2}$<br>$1.2\times10^{-1}$ | | Yaws (2003)<br>Gharagheizi et al. (2010) | X<br>Q | 237<br>246 |
| ethylpropylamine<br>$C_5H_{13}N$<br>[20193-20-8]<br>XCVNDBIXFPGMIW-UHFFFAOYSA-N | $1.7\times10^{-1}$<br>$1.6\times10^{-1}$ | | Yaws (2003)<br>Gharagheizi et al. (2010) | X<br>Q | 237<br>246 |
| ethylisopropylamine<br>$C_5H_{13}N$<br>[19961-27-4]<br>RIVIDPPYRINTTH-UHFFFAOYSA-N | $1.8\times10^{-1}$<br>$1.8\times10^{-1}$ | | Yaws (2003)<br>Gharagheizi et al. (2010) | X<br>Q | 237<br>246 |
| methyldiethylamine<br>$C_5H_{13}N$<br>[616-39-7]<br>GNVRJGIVDSQCOP-UHFFFAOYSA-N | $7.9\times10^{-2}$<br>$1.1\times10^{-1}$ | | Yaws (2003)<br>Gharagheizi et al. (2010) | X<br>Q | 237<br>246 |
| 2-methylpiperazine<br>$C_5H_{12}N_2$<br>[109-07-9]<br>JOMNTHCQHJPVAZ-UHFFFAOYSA-N | $5.0\times10^3$ | 9200 | Nguyen (2013) | M | 11 |



Table A4.1: Amines (C, H, N) (...continued)

| Substance Formula (Trivial Name) [CAS Registry Number] InChIKey | $H_s^{cp}$ (at $T^{\ominus}$) $\left[\dfrac{\text{mol}}{\text{m}^3\,\text{Pa}}\right]$ | $\dfrac{\text{d}\ln H_s^{cp}}{\text{d}(1/T)}$ [K] | Reference | Type | Note |
|---|---|---|---|---|---|
| 1-hexanamine | $3.5\times10^{-1}$ | | Brockbank (2013) | L | |
| $C_6H_{13}NH_2$ | $3.2\times10^{-1}$ | | Rytting et al. (1978) | M | |
| (hexylamine) | $3.7\times10^{-1}$ | | Christie and Crisp (1967) | M | |
| [111-26-2] | $1.0\times10^{-1}$ | | Yaws (2003) | X | 237 |
| BMVXCPBXGZKUPN-UHFFFAOYSA-N | $2.3\times10^{-1}$ | | Keshavarz et al. (2022) | Q | |
| | 2.4 | | Duchowicz et al. (2020) | Q | 184 |
| | $1.4\times10^{-1}$ | | Gharagheizi et al. (2010) | Q | 246 |
| | $3.7\times10^{-1}$ | | Hilal et al. (2008) | Q | |
| | 1.0 | | Modarresi et al. (2007) | Q | 67 |
| | $3.2\times10^{-1}$ | | Yaffe et al. (2003) | Q | 248, 249 |
| | $9.2\times10^{-2}$ | | Yao et al. (2002) | Q | 229, 267 |
| | $4.3\times10^{-1}$ | | English and Carroll (2001) | Q | 230, 274 |
| | $3.3\times10^{-1}$ | | Katritzky et al. (1998) | Q | |
| | $1.8\times10^{-1}$ | | Nirmalakhandan et al. (1997) | Q | |
| | $3.4\times10^{-1}$ | | Russell et al. (1992) | Q | 279 |
| | $4.6\times10^{-1}$ | | Suzuki et al. (1992) | Q | 232 |
| | $3.7\times10^{-1}$ | | Duchowicz et al. (2020) | ? | 185, 21 |
| | $1.0\times10^{-1}$ | | Yaws (1999) | ? | 21 |
| | $3.2\times10^{-1}$ | | Abraham et al. (1990) | ? | |
| | | 7900 | Abraham (1984) | ? | 21 |
| 2-hexanamine | $1.9\times10^{-1}$ | | Yaws (2003) | X | 237 |
| $C_6H_{15}N$ | $1.5\times10^{-1}$ | | Gharagheizi et al. (2010) | Q | 246 |
| [5329-79-3] | | | | | |
| WGBBUURBHXLGFM-UHFFFAOYSA-N | | | | | |
| 3-hexanamine | $1.8\times10^{-1}$ | | Yaws (2003) | X | 237 |
| $C_6H_{15}N$ | $1.5\times10^{-1}$ | | Gharagheizi et al. (2010) | Q | 246 |
| [16751-58-9] | | | | | |
| HQLZFBUAULNEGP-UHFFFAOYSA-N | | | | | |
| 1-amino-2-methylpentane | $2.0\times10^{-1}$ | | Yaws (2003) | X | 237 |
| $C_6H_{15}N$ | $1.8\times10^{-1}$ | | Gharagheizi et al. (2010) | Q | 246 |
| [13364-16-4] | | | | | |
| WNDXRJBYZOSNQO-UHFFFAOYSA-N | | | | | |
| 1-amino-3-methylpentane | $2.0\times10^{-1}$ | | Yaws (2003) | X | 237 |
| $C_6H_{15}N$ | $1.8\times10^{-1}$ | | Gharagheizi et al. (2010) | Q | 246 |
| [42245-37-4] | | | | | |
| JLAUIBFZZUVOBB-UHFFFAOYSA-N | | | | | |
| 1-amino-4-methylpentane | $2.0\times10^{-1}$ | | Yaws (2003) | X | 237 |
| $C_6H_{15}N$ | $1.8\times10^{-1}$ | | Gharagheizi et al. (2010) | Q | 246 |
| [5344-20-7] | | | | | |
| QVIAMKXOQGCYCV-UHFFFAOYSA-N | | | | | |
| 2-amino-3-methylpentane | $1.7\times10^{-1}$ | | Yaws (2003) | X | 237 |
| $C_6H_{15}N$ | $2.1\times10^{-1}$ | | Gharagheizi et al. (2010) | Q | 246 |
| [35399-81-6] | | | | | |
| ZFAGOADKDXXTSV-UHFFFAOYSA-N | | | | | |



Table A4.1: Amines (C, H, N) (. . . continued)

| Substance Formula (Trivial Name) [CAS Registry Number] InChIKey | $H_s^{cp}$ (at $T^{\ominus}$) $\left[\dfrac{\text{mol}}{\text{m}^3\,\text{Pa}}\right]$ | $\dfrac{\mathrm{d}\ln H_s^{cp}}{\mathrm{d}(1/T)}$ [K] | Reference | Type | Note |
|---|---|---|---|---|---|
| 2-amino-4-methylpentane C$_6$H$_{15}$N [108-09-8] UNBMPKNTYKDYCG-UHFFFAOYSA-N | $2.3\times10^{-1}$ $2.3\times10^{-1}$ $3.6\times10^{-1}$ $2.0\times10^{-1}$ $3.2\times10^{-1}$ | | Yaws (2003) Keshavarz et al. (2022) Duchowicz et al. (2020) Gharagheizi et al. (2010) Duchowicz et al. (2020) | X Q Q Q ? | 237  184 246 185, 21 |
| 3-amino-2-methylpentane C$_6$H$_{15}$N [54287-41-1] JYNQKCFJPQEXSL-UHFFFAOYSA-N | $1.7\times10^{-1}$ $2.1\times10^{-1}$ | | Yaws (2003) Gharagheizi et al. (2010) | X Q | 237 246 |
| 1-amino-2-ethylbutane C$_6$H$_{15}$N [617-79-8] MGWAGIQQTULHGU-UHFFFAOYSA-N | $1.8\times10^{-1}$ $1.8\times10^{-1}$ | | Yaws (2003) Gharagheizi et al. (2010) | X Q | 237 246 |
| 1-amino-2,2-dimethylbutane C$_6$H$_{15}$N [41781-17-3] PZVPOYBHOPRJNP-UHFFFAOYSA-N | $2.1\times10^{-1}$ $2.0\times10^{-1}$ | | Yaws (2003) Gharagheizi et al. (2010) | X Q | 237 246 |
| 1-amino-2,3-dimethylbutane C$_6$H$_{15}$N [66553-05-7] GBMSZXWHMSSBGP-UHFFFAOYSA-N | $2.5\times10^{-1}$ $2.2\times10^{-1}$ | | Yaws (2003) Gharagheizi et al. (2010) | X Q | 237 246 |
| 1-amino-3,3-dimethylbutane C$_6$H$_{15}$N [15673-00-4] GPWHFPWZAPOYNO-UHFFFAOYSA-N | $1.6\times10^{-1}$ $2.0\times10^{-1}$ | | Yaws (2003) Gharagheizi et al. (2010) | X Q | 237 246 |
| 2-amino-3,3-dimethylbutane C$_6$H$_{15}$N [3850-30-4] DXSUORGKJZADET-UHFFFAOYSA-N | $2.4\times10^{-1}$ $2.4\times10^{-1}$ | | Yaws (2003) Gharagheizi et al. (2010) | X Q | 237 246 |
| dimethylbutylamine C$_6$H$_{15}$N [927-62-8] DJEQZVQFEPKLOY-UHFFFAOYSA-N | $8.9\times10^{-2}$ $4.8\times10^{-2}$ $6.7\times10^{-2}$ | | Yaws (2003) Gharagheizi et al. (2012) Gharagheizi et al. (2010) | X Q Q | 237  246 |
| dimethyl-*sec*-butylamine C$_6$H$_{15}$N [921-04-0] USSPHSVODLAWSA-UHFFFAOYSA-N | $1.0\times10^{-1}$ $1.2\times10^{-1}$ $8.8\times10^{-2}$ | | Yaws (2003) Gharagheizi et al. (2012) Gharagheizi et al. (2010) | X Q Q | 237  246 |
| dimethyl-*tert*-butylamine C$_6$H$_{15}$N [918-02-5] OXQMIXBVXHWDPX-UHFFFAOYSA-N | $1.0\times10^{-1}$ $9.0\times10^{-2}$ | | Yaws (2003) Gharagheizi et al. (2010) | X Q | 237 246 |



Table A4.1: Amines (C, H, N) (... continued)

| Substance Formula (Trivial Name) [CAS Registry Number] InChIKey | $H_s^{cp}$ (at $T^{\ominus}$) $\left[\dfrac{\mathrm{mol}}{\mathrm{m^3\,Pa}}\right]$ | $\dfrac{\mathrm{d}\ln H_s^{cp}}{\mathrm{d}(1/T)}$ [K] | Reference | Type | Note |
|---|---|---|---|---|---|
| ethylbutylamine $C_6H_{15}N$ [13360-63-9] QHCCDDQKNUYGNC-UHFFFAOYSA-N | $1.3\times10^{-1}$ $1.2\times10^{-1}$ | | Yaws (2003) Gharagheizi et al. (2010) | X Q | 237 246 |
| ethylisobutylamine $C_6H_{15}N$ [13205-60-2] FNLUJDLKYOWMMF-UHFFFAOYSA-N | $1.5\times10^{-1}$ $1.2\times10^{-1}$ | | Yaws (2003) Gharagheizi et al. (2010) | X Q | 237 246 |
| ethyl-*sec*-butylamine $C_6H_{15}N$ [21035-44-9] KFYKZKISJBGVMR-UHFFFAOYSA-N | $1.5\times10^{-1}$ $1.3\times10^{-1}$ | | Yaws (2003) Gharagheizi et al. (2010) | X Q | 237 246 |
| methyl-1,1-dimethylpropylamine $C_6H_{15}N$ [2978-64-5] BUJFTKPQXSIZFX-UHFFFAOYSA-N | $1.1\times10^{-1}$ $1.8\times10^{-1}$ $1.4\times10^{-1}$ | | Yaws (2003) Gharagheizi et al. (2012) Gharagheizi et al. (2010) | X Q Q | 237 246 |
| methyl-1,2-dimethylpropylamine $C_6H_{15}N$ [34317-39-0] LJLWVVCWBURGCC-UHFFFAOYSA-N | $1.2\times10^{-1}$ $5.9\times10^{-2}$ $1.4\times10^{-1}$ | | Yaws (2003) Gharagheizi et al. (2012) Gharagheizi et al. (2010) | X Q Q | 237 246 |
| methyl-2,2-dimethylpropylamine $C_6H_{15}N$ [26153-91-3] UQGXHNDRCRTZAC-UHFFFAOYSA-N | $1.1\times10^{-1}$ $1.1\times10^{-1}$ | | Yaws (2003) Gharagheizi et al. (2010) | X Q | 237 246 |
| methyl-1-methylbutylamine $C_6H_{15}N$ [51932-19-5] IPBXLJFBVNLKFE-UHFFFAOYSA-N | $1.2\times10^{-1}$ $8.7\times10^{-2}$ $1.2\times10^{-1}$ | | Yaws (2003) Gharagheizi et al. (2012) Gharagheizi et al. (2010) | X Q Q | 237 246 |
| methylethylpropylamine $C_6H_{15}N$ [4458-32-6] SMBYUOXUISCLCF-UHFFFAOYSA-N | $1.1\times10^{-1}$ $1.0\times10^{-1}$ $8.2\times10^{-2}$ | | Yaws (2003) Gharagheizi et al. (2012) Gharagheizi et al. (2010) | X Q Q | 237 246 |
| methylethylisopropylamine $C_6H_{15}N$ [39198-07-7] UTLDDSNRFHWERZ-UHFFFAOYSA-N | $1.1\times10^{-1}$ $9.8\times10^{-2}$ | | Yaws (2003) Gharagheizi et al. (2010) | X Q | 237 246 |
| methylpentylamine $C_6H_{15}N$ [25419-06-1] UOIWOHLIGKIYFE-UHFFFAOYSA-N | $1.2\times10^{-1}$ $1.1\times10^{-1}$ | | Yaws (2003) Gharagheizi et al. (2010) | X Q | 237 246 |





Table A4.1: Amines (C, H, N) (...continued)

| Substance Formula (Trivial Name) [CAS Registry Number] InChIKey | $H_s^{cp}$ (at $T^{\ominus}$) $\left[\dfrac{\mathrm{mol}}{\mathrm{m^3\,Pa}}\right]$ | $\dfrac{\mathrm{d}\ln H_s^{cp}}{\mathrm{d}(1/T)}$ [K] | Reference | Type | Note |
|---|---|---|---|---|---|
| propylisopropylamine $C_6H_{15}N$ [21968-17-2] VLSTXUUYLIALPB-UHFFFAOYSA-N | $1.4\times10^{-1}$ $1.3\times10^{-1}$ | | Yaws (2003) Gharagheizi et al. (2010) | X Q | 237 246 |
| 1,6-hexanediamine $C_6H_{16}N_2$ (hexamethylenediamine) [124-09-4] NAQMVNRVTILPCV-UHFFFAOYSA-N | $1.0\times10^{3}$ $3.1\times10^{3}$ | 5000 | Nguyen (2013) HSDB (2015) | M Q | 11 99 |
| N,N'-methanetetraylbis-2-propanamine $C_7H_{14}N_2$ (1,3-diisopropylcarbodiimide) [693-13-0] BDNKZNFMNDZQMI-UHFFFAOYSA-N | $9.9\times10^{-3}$ | | HSDB (2015) | Q | 447 |
| 4-methyl-2-hexanamine $C_7H_{17}N$ [105-41-9] YAHRDLICUYEDAU-UHFFFAOYSA-N | $2.3\times10^{-1}$ | | HSDB (2015) | Q | 447 |
| 1-heptanamine $C_7H_{17}N$ (1-heptylamine) [111-68-2] WJYIASZWHGOTOU-UHFFFAOYSA-N | $2.8\times10^{-1}$ $2.4\times10^{-1}$ $8.0\times10^{-2}$ $8.3\times10^{-2}$ $4.5\times10^{-1}$ $9.6\times10^{-1}$ $2.1\times10^{-1}$ $3.3\times10^{-1}$ $1.4\times10^{-1}$ $2.4\times10^{-1}$ | | Brockbank (2013) Rytting et al. (1978) Yaws (2003) Gharagheizi et al. (2010) Hilal et al. (2008) Modarresi et al. (2007) Yaffe et al. (2003) English and Carroll (2001) Nirmalakhandan et al. (1997) Abraham et al. (1990) | L M X Q Q Q Q Q Q ? | 237 246 67 248, 272 230, 231 |
| diethylisopropylamine $C_7H_{17}N$ [6006-15-1] ULWOJODHECIZAU-UHFFFAOYSA-N | $7.7\times10^{-2}$ $8.0\times10^{-2}$ | | Yaws (2003) Gharagheizi et al. (2010) | X Q | 237 246 |
| diethylpropylamine $C_7H_{17}N$ [4458-31-5] PQZTVWVYCLIIJY-UHFFFAOYSA-N | $7.7\times10^{-2}$ $1.5\times10^{-1}$ $7.3\times10^{-2}$ | | Yaws (2003) Gharagheizi et al. (2012) Gharagheizi et al. (2010) | X Q Q | 237 246 |
| dimethyl-1,1-dimethylpropylamine $C_7H_{17}N$ [57757-60-5] CUHMMDPUXJFCNB-UHFFFAOYSA-N | $6.3\times10^{-2}$ $6.5\times10^{-2}$ | | Yaws (2003) Gharagheizi et al. (2010) | X Q | 237 246 |



Table A4.1: Amines (C, H, N) (... continued)

| Substance<br>Formula<br>(Trivial Name)<br>[CAS Registry Number]<br>InChIKey | $H_s^{cp}$<br>(at $T^{\ominus}$)<br>$\left[\dfrac{\text{mol}}{\text{m}^3\,\text{Pa}}\right]$ | $\dfrac{\text{d}\ln H_s^{cp}}{\text{d}(1/T)}$<br><br>[K] | Reference | Type | Note |
|---|---|---|---|---|---|
| dimethyl-1,2-dimethylpropylamine<br>C$_7$H$_{17}$N<br>[66225-38-5]<br>FWBCYOHCOBOARU-UHFFFAOYSA-N | $6.9\times10^{-2}$<br>$5.7\times10^{-2}$<br>$7.8\times10^{-2}$ | | Yaws (2003)<br>Gharagheizi et al. (2012)<br>Gharagheizi et al. (2010) | X<br>Q<br>Q | 237<br><br>246 |
| dimethyl-2,2-dimethylpropylamine<br>C$_7$H$_{17}$N<br>[10076-31-0]<br>FUIRUFXAVIHAQB-UHFFFAOYSA-N | $6.3\times10^{-2}$<br>$1.8\times10^{-2}$<br>$5.8\times10^{-2}$ | | Yaws (2003)<br>Gharagheizi et al. (2012)<br>Gharagheizi et al. (2010) | X<br>Q<br>Q | 237<br><br>246 |
| dimethyl-2-methylbutylamine<br>C$_7$H$_{17}$N<br>[66225-39-6]<br>BHMZPPHMQJHCHQ-UHFFFAOYSA-N | $7.1\times10^{-2}$<br>$2.4\times10^{-2}$<br>$6.0\times10^{-2}$ | | Yaws (2003)<br>Gharagheizi et al. (2012)<br>Gharagheizi et al. (2010) | X<br>Q<br>Q | 237<br><br>246 |
| dimethyl-3-methylbutylamine<br>C$_7$H$_{17}$N<br>[2315-43-7]<br>KOOQJINBDNZUTB-UHFFFAOYSA-N | $7.2\times10^{-2}$<br>$3.5\times10^{-2}$<br>$5.6\times10^{-2}$ | | Yaws (2003)<br>Gharagheizi et al. (2012)<br>Gharagheizi et al. (2010) | X<br>Q<br>Q | 237<br><br>246 |
| dimethylpentylamine<br>C$_7$H$_{17}$N<br>[26153-88-8]<br>IDFANOPDMXWIOP-UHFFFAOYSA-N | $5.3\times10^{-2}$<br>$4.1\times10^{-2}$<br>$5.0\times10^{-2}$ | | Yaws (2003)<br>Gharagheizi et al. (2012)<br>Gharagheizi et al. (2010) | X<br>Q<br>Q | 237<br><br>246 |
| dimethyl-2-pentylamine<br>C$_7$H$_{17}$N<br>[57303-85-2]<br>LSTZYJQJHGEVKH-UHFFFAOYSA-N | $7.7\times10^{-2}$<br>$6.6\times10^{-2}$<br>$6.0\times10^{-2}$ | | Yaws (2003)<br>Gharagheizi et al. (2012)<br>Gharagheizi et al. (2010) | X<br>Q<br>Q | 237<br><br>246 |
| dimethyl-3-pentylamine<br>C$_7$H$_{17}$N<br>[18636-94-7]<br>SUEKSPIQGIMSQM-UHFFFAOYSA-N | $7.0\times10^{-2}$<br>$5.5\times10^{-2}$<br>$6.5\times10^{-2}$ | | Yaws (2003)<br>Gharagheizi et al. (2012)<br>Gharagheizi et al. (2010) | X<br>Q<br>Q | 237<br><br>246 |
| ethylpentylamine<br>C$_7$H$_{17}$N<br>[17839-26-8]<br>ICVFPLUSMYSIFO-UHFFFAOYSA-N | $8.6\times10^{-2}$<br>$5.6\times10^{-2}$<br>$9.1\times10^{-2}$ | | Yaws (2003)<br>Gharagheizi et al. (2012)<br>Gharagheizi et al. (2010) | X<br>Q<br>Q | 237<br><br>246 |
| methyldiisopropylamine<br>C$_7$H$_{17}$N<br>[10342-97-9]<br>ISRXMEYARGEVIU-UHFFFAOYSA-N | $7.1\times10^{-2}$<br>$7.5\times10^{-2}$ | | Yaws (2003)<br>Gharagheizi et al. (2010) | X<br>Q | 237<br>246 |
| methyldipropylamine<br>C$_7$H$_{17}$N<br>[3405-42-3]<br>UVBMZKBIZUWTLV-UHFFFAOYSA-N | $7.5\times10^{-2}$<br>$5.1\times10^{-2}$<br>$6.4\times10^{-2}$ | | Yaws (2003)<br>Gharagheizi et al. (2012)<br>Gharagheizi et al. (2010) | X<br>Q<br>Q | 237<br><br>246 |





Table A4.1: Amines (C, H, N) (...continued)

| Substance Formula (Trivial Name) [CAS Registry Number] InChIKey | $H_s^{cp}$ (at $T^{\ominus}$) $\left[\dfrac{\text{mol}}{\text{m}^3\,\text{Pa}}\right]$ | $\dfrac{\text{d}\ln H_s^{cp}}{\text{d}(1/T)}$ [K] | Reference | Type | Note |
|---|---|---|---|---|---|
| methylethylbutylamine $C_7H_{17}N$ [66225-40-9] WOLFCKKMHUVEPN-UHFFFAOYSA-N | $7.2\times10^{-2}$ $8.2\times10^{-2}$ $6.0\times10^{-2}$ | | Yaws (2003) Gharagheizi et al. (2012) Gharagheizi et al. (2010) | X Q Q | 237 246 |
| methylethylisobutylamine $C_7H_{17}N$ [60247-14-5] QQWXQKMVASVXCI-UHFFFAOYSA-N | $7.8\times10^{-2}$ $4.3\times10^{-2}$ $6.9\times10^{-2}$ | | Yaws (2003) Gharagheizi et al. (2012) Gharagheizi et al. (2010) | X Q Q | 237 246 |
| methylethyl-*sec*-butylamine $C_7H_{17}N$ [66225-41-0] HAFZSBASGRZPLA-UHFFFAOYSA-N | $8.2\times10^{-2}$ $7.1\times10^{-2}$ | | Yaws (2003) Gharagheizi et al. (2010) | X Q | 237 246 |
| methylethyl-*tert*-butylamine $C_7H_{17}N$ [52841-28-8] BWWLCYLGZIWOHK-UHFFFAOYSA-N | $7.0\times10^{-2}$ $7.6\times10^{-2}$ | | Yaws (2003) Gharagheizi et al. (2010) | X Q | 237 246 |
| methylhexylamine $C_7H_{17}N$ [35161-70-7] XJINZNWPEQMMBV-UHFFFAOYSA-N | $8.3\times10^{-2}$ $3.5\times10^{-2}$ $8.6\times10^{-2}$ | | Yaws (2003) Gharagheizi et al. (2012) Gharagheizi et al. (2010) | X Q Q | 237 246 |
| methylpropylisopropylamine $C_7H_{17}N$ [66225-42-1] OYQDUCLFZSKBCZ-UHFFFAOYSA-N | $7.3\times10^{-2}$ $1.6\times10^{-1}$ $7.1\times10^{-2}$ | | Yaws (2003) Gharagheizi et al. (2012) Gharagheizi et al. (2010) | X Q Q | 237 246 |
| 1-octanamine $C_8H19N$ (octylamine) [111-86-4] IOQPZZOEVPZRBK-UHFFFAOYSA-N | $2.2\times10^{-1}$ $1.9\times10^{-1}$ $1.2\times10^{-2}$ 2.4 $1.9\times10^{-2}$ $4.3\times10^{-1}$ $2.1\times10^{-1}$ $2.5\times10^{-1}$ $1.1\times10^{-1}$ $1.2\times10^{-2}$ $1.9\times10^{-1}$ | 7400 6600 | Brockbank (2013) Rytting et al. (1978) Duchowicz et al. (2020) Duchowicz et al. (2020) Gharagheizi et al. (2012) Hilal et al. (2008) Kühne et al. (2005) Yaffe et al. (2003) English and Carroll (2001) Nirmalakhandan et al. (1997) Kühne et al. (2005) Yaws (1999) Abraham et al. (1990) | L M V Q Q Q Q Q Q Q ? ? ? | 186 248, 249 230, 231 21 |
| 2-ethyl-1-hexanamine $C_8H_{19}N$ (2-ethylhexylamine) [104-75-6] LTHNHFOGQMKPOV-UHFFFAOYSA-N | $1.0\times10^{-1}$ $9.4\times10^{-1}$ $3.7\times10^{-1}$ $9.7\times10^{-1}$ | 7400 7400 | Duchowicz et al. (2020) Duchowicz et al. (2020) Hilal et al. (2008) Modarresi et al. (2007) Kühne et al. (2005) Kühne et al. (2005) | V Q Q Q Q ? | 186 67 |



Table A4.1: Amines (C, H, N) (... continued)

| Substance<br>Formula<br>(Trivial Name)<br>[CAS Registry Number]<br>InChIKey | $H_s^{cp}$<br>(at $T^\ominus$)<br><br>$\left[\dfrac{\mathrm{mol}}{\mathrm{m^3\,Pa}}\right]$ | $\dfrac{\mathrm{d}\ln H_s^{cp}}{\mathrm{d}(1/T)}$<br><br>[K] | Reference | Type | Note |
|---|---|---|---|---|---|
| diethylbutylamine<br>$C_8H_{19}N$<br>[4444-68-2]<br>ORSUTASIQKBEFU-UHFFFAOYSA-N | $5.2\times10^{-2}$<br>$1.2\times10^{-1}$<br>$5.3\times10^{-2}$ | | Yaws (2003)<br>Gharagheizi et al. (2012)<br>Gharagheizi et al. (2010) | X<br>Q<br>Q | 237<br><br>246 |
| dimethylhexylamine<br>$C_8H_{19}N$<br>[4385-04-0]<br>QMHNQZGXPNCMCO-UHFFFAOYSA-N | $3.4\times10^{-2}$<br>$3.1\times10^{-2}$<br>$3.8\times10^{-2}$ | | Yaws (2003)<br>Gharagheizi et al. (2012)<br>Gharagheizi et al. (2010) | X<br>Q<br>Q | 237<br><br>246 |
| ethylhexylamine<br>$C_8H_{19}N$<br>[20352-67-4]<br>WSTNFGAKGUERTC-UHFFFAOYSA-N | $6.2\times10^{-2}$<br>$4.5\times10^{-2}$<br>$7.3\times10^{-2}$ | | Yaws (2003)<br>Gharagheizi et al. (2012)<br>Gharagheizi et al. (2010) | X<br>Q<br>Q | 237<br><br>246 |
| methylheptylamine<br>$C_8H_{19}N$<br>[36343-05-2]<br>LTGYRKOQQQWWAF-UHFFFAOYSA-N | $6.2\times10^{-2}$<br>$2.6\times10^{-2}$<br>$7.1\times10^{-2}$ | | Yaws (2003)<br>Gharagheizi et al. (2012)<br>Gharagheizi et al. (2010) | X<br>Q<br>Q | 237<br><br>246 |
| 2,2'-azobis-(2-methylpropanenitrile)<br>$C_8H_{12}N_4$<br>[78-67-1]<br>OZAIFHULBGXAKX-UHFFFAOYSA-N | 2.4<br><br>$8.0\times10^{1}$ | | Duchowicz et al. (2020)<br><br>Duchowicz et al. (2020) | V<br><br>Q | 186 |
| diethylpentylamine<br>$C_9H_{21}N$<br>[2162-91-6]<br>YZULHOOBWDXEOT-UHFFFAOYSA-N | $4.0\times10^{-2}$<br>$8.6\times10^{-2}$<br>$4.0\times10^{-2}$ | | Yaws (2003)<br>Gharagheizi et al. (2012)<br>Gharagheizi et al. (2010) | X<br>Q<br>Q | 237<br><br>246 |
| dimethylheptylamine<br>$C_9H_{21}N$<br>[5277-11-2]<br>LSICDRUYCNGRIF-UHFFFAOYSA-N | $2.5\times10^{-2}$<br>$2.7\times10^{-2}$<br>$3.0\times10^{-2}$ | | Yaws (2003)<br>Gharagheizi et al. (2012)<br>Gharagheizi et al. (2010) | X<br>Q<br>Q | 237<br><br>246 |
| ethylheptylamine<br>$C_9H_{21}N$<br>[66793-76-8]<br>IUZZLNVABCISOI-UHFFFAOYSA-N | $5.2\times10^{-2}$<br>$5.3\times10^{-2}$<br>$6.3\times10^{-2}$ | | Yaws (2003)<br>Gharagheizi et al. (2012)<br>Gharagheizi et al. (2010) | X<br>Q<br>Q | 237<br><br>246 |
| methyloctylamine<br>$C_9H_{21}N$<br>[2439-54-5]<br>SEGJNMCIMOLEDM-UHFFFAOYSA-N | $5.4\times10^{-2}$<br>$3.0\times10^{-2}$<br>$6.2\times10^{-2}$ | | Yaws (2003)<br>Gharagheizi et al. (2012)<br>Gharagheizi et al. (2010) | X<br>Q<br>Q | 237<br><br>246 |
| nonylamine<br>$C_9H_{21}N$<br>[112-20-9]<br>FJDUDHYHRVPMJZ-UHFFFAOYSA-N | $5.9\times10^{-2}$<br>$2.8\times10^{-2}$ | | Yaws (2003)<br>Gharagheizi et al. (2010) | X<br>Q | 237<br>246 |



Table A4.1: Amines (C, H, N) (. . . continued)

| Substance Formula (Trivial Name) [CAS Registry Number] InChIKey | $H_s^{cp}$ (at $T^{\ominus}$) $\left[\dfrac{\mathrm{mol}}{\mathrm{m^3\,Pa}}\right]$ | $\dfrac{\mathrm{d}\ln H_s^{cp}}{\mathrm{d}(1/T)}$ [K] | Reference | Type | Note |
|---|---|---|---|---|---|
| diethylhexylamine $C_{10}H_{23}N$ [44979-90-0] XHDKYWMKOLURNK-UHFFFAOYSA-N | $3.3\times10^{-2}$ $6.7\times10^{-2}$ $3.2\times10^{-2}$ | | Yaws (2003) Gharagheizi et al. (2012) Gharagheizi et al. (2010) | X Q Q | 237 246 |
| dimethyloctylamine $C_{10}H_{23}N$ [7378-99-6] UQKAOOAFEFCDGT-UHFFFAOYSA-N | $2.0\times10^{-2}$ $4.5\times10^{-2}$ $2.5\times10^{-2}$ | | Yaws (2003) Gharagheizi et al. (2012) Gharagheizi et al. (2010) | X Q Q | 237 246 |
| dipentylamine $C_{10}H_{23}N$ [2050-92-2] JACMPVXHEARCBO-UHFFFAOYSA-N | $5.3\times10^{-2}$ $4.4\times10^{-2}$ $6.4\times10^{-2}$ | | Yaws (2003) Gharagheizi et al. (2012) Gharagheizi et al. (2010) | X Q Q | 237 246 |
| ethyloctylamine $C_{10}H_{23}N$ [4088-36-2] SDQCOADWEMMSGK-UHFFFAOYSA-N | $5.2\times10^{-2}$ $6.2\times10^{-2}$ $6.0\times10^{-2}$ | | Yaws (2003) Gharagheizi et al. (2012) Gharagheizi et al. (2010) | X Q Q | 237 246 |
| methylnonylamine $C_{10}H_{23}N$ [39093-27-1] OZIXTIPURXIEMB-UHFFFAOYSA-N | $5.6\times10^{-2}$ $2.2\times10^{-2}$ $6.1\times10^{-2}$ | | Yaws (2003) Gharagheizi et al. (2012) Gharagheizi et al. (2010) | X Q Q | 237 246 |
| diethylheptylamine $C_{11}H_{25}N$ [26981-81-7] YUCNJMBRLIZNMO-UHFFFAOYSA-N | $2.3\times10^{-2}$ $1.1\times10^{-1}$ $2.7\times10^{-2}$ | | Yaws (2003) Gharagheizi et al. (2012) Gharagheizi et al. (2010) | X Q Q | 237 246 |
| dimethylnonylamine $C_{11}H_{25}N$ [17373-27-2] AMAADDMFZSZCNT-UHFFFAOYSA-N | $1.5\times10^{-2}$ $3.7\times10^{-2}$ $2.2\times10^{-2}$ | | Yaws (2003) Gharagheizi et al. (2012) Gharagheizi et al. (2010) | X Q Q | 237 246 |
| diethyloctylamine $C_{12}H_{27}N$ [4088-37-3] BVUGARXRRGZONH-UHFFFAOYSA-N | $2.3\times10^{-2}$ $2.6\times10^{-2}$ | | Yaws (2003) Gharagheizi et al. (2010) | X Q | 237 246 |
| dihexylamine $C_{12}H_{27}N$ [143-16-8] PXSXRABJBXYMFT-UHFFFAOYSA-N | $1.3\times10^{-1}$ $8.9\times10^{-2}$ | | Yaws (2003) Gharagheizi et al. (2010) | X Q | 237 246 |
| dimethyldecylamine $C_{12}H_{27}N$ [1120-24-7] YWWNNLPSZSEZNZ-UHFFFAOYSA-N | $1.6\times10^{-2}$ $4.3\times10^{-2}$ $2.2\times10^{-2}$ | | Yaws (2003) Gharagheizi et al. (2012) Gharagheizi et al. (2010) | X Q Q | 237 246 |





Table A4.1: Amines (C, H, N) (. . . continued)

| Substance<br>Formula<br>(Trivial Name)<br>[CAS Registry Number]<br>InChIKey | $H_s^{cp}$<br>(at $T^{\ominus}$)<br>$\left[\dfrac{\mathrm{mol}}{\mathrm{m^3\,Pa}}\right]$ | $\dfrac{\mathrm{d}\ln H_s^{cp}}{\mathrm{d}(1/T)}$<br><br>[K] | Reference | Type | Note |
|---|---|---|---|---|---|
| methylundecylamine<br>$C_{12}H_{27}N$<br>[66553-53-5]<br>JCJFBKCQLFMABE-UHFFFAOYSA-N | $1.3\times10^{-1}$<br>$8.8\times10^{-2}$ | | Yaws (2003)<br>Gharagheizi et al. (2010) | X<br>Q | 237<br>246 |
| diethylnonylamine<br>$C_{13}H_{29}N$<br>[45124-35-4]<br>IBTOMDSHMLGUHA-UHFFFAOYSA-N | $3.1\times10^{-2}$<br>$1.2\times10^{-1}$<br>$2.8\times10^{-2}$ | | Yaws (2003)<br>Gharagheizi et al. (2012)<br>Gharagheizi et al. (2010) | X<br>Q<br>Q | 237<br><br>246 |
| dimethylundecylamine<br>$C_{13}H_{29}N$<br>[17373-28-3]<br>MMWFTWUMBYZIRZ-UHFFFAOYSA-N | $2.7\times10^{-2}$<br>$4.6\times10^{-2}$<br>$2.6\times10^{-2}$ | | Yaws (2003)<br>Gharagheizi et al. (2012)<br>Gharagheizi et al. (2010) | X<br>Q<br>Q | 237<br><br>246 |
| ethylundecylamine<br>$C_{13}H_{29}N$<br>[59570-04-6]<br>LKVBHKWFYHKTSM-UHFFFAOYSA-N | $1.4\times10^{-1}$<br>$1.2\times10^{-1}$ | | Yaws (2003)<br>Gharagheizi et al. (2010) | X<br>Q | 237<br>246 |
| methyldodecylamine<br>$C_{13}H_{29}N$<br>[7311-30-0]<br>OMEMQVZNTDHENJ-UHFFFAOYSA-N | $1.5\times10^{-1}$<br>$1.4\times10^{-1}$ | | Yaws (2003)<br>Gharagheizi et al. (2010) | X<br>Q | 237<br>246 |
| diethyldecylamine<br>$C_{14}H_{31}N$<br>[6308-94-7]<br>UFFQZCPLBHYOFV-UHFFFAOYSA-N | $4.2\times10^{-2}$<br>$3.5\times10^{-2}$ | | Yaws (2003)<br>Gharagheizi et al. (2010) | X<br>Q | 237<br>246 |
| diheptylamine<br>$C_{14}H_{31}N$<br>[2470-68-0]<br>NJWMENBYMFZACG-UHFFFAOYSA-N | $1.7\times10^{-1}$<br>$2.7\times10^{-1}$ | | Yaws (2003)<br>Gharagheizi et al. (2010) | X<br>Q | 237<br>246 |
| ethyldodecylamine<br>$C_{14}H_{31}N$<br>[35902-57-9]<br>LWIPGCTWFZCIKX-UHFFFAOYSA-N | $1.8\times10^{-1}$<br>$2.3\times10^{-1}$ | | Yaws (2003)<br>Gharagheizi et al. (2010) | X<br>Q | 237<br>246 |
| methyltridecylamine<br>$C_{14}H_{31}N$<br>[45165-81-9]<br>XMRPIOZXPHTSCE-UHFFFAOYSA-N | $2.0\times10^{-1}$<br>$2.8\times10^{-1}$ | | Yaws (2003)<br>Gharagheizi et al. (2010) | X<br>Q | 237<br>246 |
| tripentylamine<br>$C_{15}H_{33}N$<br>[621-77-2]<br>OOHAUGDGCWURIT-UHFFFAOYSA-N | $6.7\times10^{-2}$<br>$3.1\times10^{-2}$<br>$7.1\times10^{-2}$ | | Yaws (2003)<br>Gharagheizi et al. (2012)<br>Gharagheizi et al. (2010) | X<br>Q<br>Q | 237<br><br>246 |





Table A4.1: Amines (C, H, N) (... continued)

| Substance Formula (Trivial Name) [CAS Registry Number] InChIKey | $H_s^{cp}$ (at $T^\ominus$) $\left[\dfrac{\text{mol}}{\text{m}^3\,\text{Pa}}\right]$ | $\dfrac{\text{d}\ln H_s^{cp}}{\text{d}(1/T)}$ [K] | Reference | Type | Note |
|---|---|---|---|---|---|
| 1-tridecanamine $C_{13}H_{29}N$ [2869-34-3] ABVVEAHYODGCLZ-UHFFFAOYSA-N | $9.0\times10^{-2}$ | | Altschuh et al. (1999) | M | |
| dimethylamine $(CH_3)_2NH$ [124-40-3] ROSDSFDQCJNGOL-UHFFFAOYSA-N | $5.5\times10^{-1}$ | | Burkholder et al. (2019) | L | |
| | $5.5\times10^{-1}$ | | Burkholder et al. (2015) | L | |
| | $6.2\times10^{-1}$ | | Brockbank (2013) | L | |
| | $3.0\times10^{-1}$ | 4000 | Wilhelm et al. (1977) | L | |
| | $5.6\times10^{-1}$ | | Christie and Crisp (1967) | M | |
| | $5.8\times10^{-1}$ | 6400 | Bagno et al. (1991) | T | 473 |
| | $6.8\times10^{-2}$ | | Keshavarz et al. (2022) | Q | |
| | 1.8 | | Duchowicz et al. (2020) | Q | 184 |
| | $5.6\times10^{-1}$ | | Li et al. (2014) | Q | 241 |
| | $6.0\times10^{-1}$ | | Hilal et al. (2008) | Q | |
| | $8.0\times10^{-1}$ | | Modarresi et al. (2007) | Q | 67 |
| | $6.7\times10^{-1}$ | | Yaffe et al. (2003) | Q | 248, 272 |
| | $4.7\times10^{-1}$ | | Katritzky et al. (1998) | Q | |
| | $5.4\times10^{-1}$ | | Nirmalakhandan et al. (1997) | Q | |
| | $3.7\times10^{-1}$ | | Russell et al. (1992) | Q | 279 |
| | $8.0\times10^{-1}$ | | Suzuki et al. (1992) | Q | 403, 232 |
| | $5.6\times10^{-1}$ | | Duchowicz et al. (2020) | ? | 185, 21 |
| | $5.6\times10^{-1}$ | | Mackay et al. (2006d) | ? | |
| | $5.7\times10^{-1}$ | | Abraham et al. (1990) | ? | |
| diethylamine $(C_2H_5)_2NH$ [109-89-7] HPNMFZURTQLUMO-UHFFFAOYSA-N | $3.8\times10^{-1}$ | | Burkholder et al. (2019) | L | |
| | $3.8\times10^{-1}$ | | Burkholder et al. (2015) | L | |
| | $3.7\times10^{-1}$ | | Brockbank (2013) | L | |
| | $3.9\times10^{-1}$ | | Christie and Crisp (1967) | M | |
| | $4.1\times10^{-1}$ | 7700 | Bagno et al. (1991) | T | 473 |
| | $2.9\times10^{-1}$ | | Yaws (2003) | X | 258 |
| | 1.3 | 10000 | Goldstein (1982) | X | 298 |
| | $5.2\times10^{-2}$ | | Dupeux et al. (2022) | Q | 259 |
| | $1.2\times10^{-1}$ | | Keshavarz et al. (2022) | Q | |
| | $2.1\times10^{-1}$ | | Duchowicz et al. (2020) | Q | |
| | $1.8\times10^{-1}$ | | Hilal et al. (2008) | Q | |
| | $2.5\times10^{-1}$ | | Modarresi et al. (2007) | Q | 67 |
| | $5.2\times10^{-1}$ | | Yaffe et al. (2003) | Q | 248, 272 |
| | $2.5\times10^{-1}$ | | Yao et al. (2002) | Q | 229 |
| | $3.1\times10^{-1}$ | | English and Carroll (2001) | Q | 230, 231 |
| | $1.5\times10^{-1}$ | | Katritzky et al. (1998) | Q | |
| | $1.5\times10^{-1}$ | | Russell et al. (1992) | Q | 358 |
| | $4.7\times10^{-1}$ | | Suzuki et al. (1992) | Q | 232 |
| | $3.9\times10^{-1}$ | | Duchowicz et al. (2020) | ? | 185, 21 |
| | $3.8\times10^{-1}$ | | Mackay et al. (2006d) | ? | |
| | $1.4\times10^{-1}$ | | Yaws (1999) | ? | 21 |
| | $1.5\times10^{-1}$ | | Yaws and Yang (1992) | ? | 21 |
| | $3.9\times10^{-1}$ | | Abraham et al. (1990) | ? | |



Table A4.1: Amines (C, H, N) (...continued)

| Substance Formula (Trivial Name) [CAS Registry Number] InChIKey | $H_s^{cp}$ (at $T^{\ominus}$) $\left[\dfrac{\mathrm{mol}}{\mathrm{m^3\,Pa}}\right]$ | $\dfrac{\mathrm{d}\ln H_s^{cp}}{\mathrm{d}(1/T)}$ [K] | Reference | Type | Note |
|---|---|---|---|---|---|
| dipropylamine $(C_3H_7)_2NH$ [142-84-7] WEHWNAOGRSTTBQ-UHFFFAOYSA-N | $2.0\times10^{-1}$ | | Burkholder et al. (2019) | L | |
| | $2.0\times10^{-1}$ | | Burkholder et al. (2015) | L | |
| | $2.1\times10^{-1}$ | | Brockbank (2013) | L | |
| | $1.2\times10^{-1}$ | 8900 | Leng et al. (2015a) | M | |
| | $1.9\times10^{-1}$ | | Christie and Crisp (1967) | M | |
| | $2.3\times10^{-1}$ | | Keshavarz et al. (2022) | Q | |
| | $2.2\times10^{-1}$ | | Duchowicz et al. (2020) | Q | |
| | $7.0\times10^{-2}$ | | Gharagheizi et al. (2012) | Q | |
| | $1.1\times10^{-1}$ | | Hilal et al. (2008) | Q | |
| | $2.7\times10^{-1}$ | | Modarresi et al. (2007) | Q | 67 |
| | | 6900 | Kühne et al. (2005) | Q | |
| | $2.1\times10^{-1}$ | | Yaffe et al. (2003) | Q | 248, 249 |
| | $9.6\times10^{-2}$ | | Yao et al. (2002) | Q | 229 |
| | $1.9\times10^{-1}$ | | English and Carroll (2001) | Q | 230, 260 |
| | $1.1\times10^{-1}$ | | Katritzky et al. (1998) | Q | |
| | $2.3\times10^{-1}$ | | Nirmalakhandan et al. (1997) | Q | |
| | $2.8\times10^{-1}$ | | Suzuki et al. (1992) | Q | 232 |
| | $1.9\times10^{-1}$ | | Duchowicz et al. (2020) | ? | 185, 21 |
| | | 8100 | Kühne et al. (2005) | ? | |
| | $1.1\times10^{-1}$ | | Yaws (1999) | ? | 21, 551 |
| | $1.9\times10^{-1}$ | | Abraham et al. (1990) | ? | |
| | | | Betterton (1992) | W | 552 |
| N-methylpropanamine $C_4H_{11}N$ [627-35-0] GVWISOJSERXQBM-UHFFFAOYSA-N | $1.9\times10^{-1}$ | | Hilal et al. (2008) | Q | |
| N-methyl-2-propanamine $C_4H_{11}N$ [4747-21-1] XHFGWHUWQXTGAT-UHFFFAOYSA-N | $1.6\times10^{-1}$ | | Gharagheizi et al. (2012) | Q | |
| | $1.4\times10^{-1}$ | | Hilal et al. (2008) | Q | |
| N-(1-methylethyl)-2-propanamine $C_6H_{15}N$ (diisopropylamine) [108-18-9] UAOMVDZJSHZZME-UHFFFAOYSA-N | $1.0\times10^{-1}$ | | Duchowicz et al. (2020) | V | 186 |
| | $1.3\times10^{-1}$ | | Yaws (2003) | X | 237 |
| | $3.3\times10^{-2}$ | | Duchowicz et al. (2020) | Q | |
| | $1.4\times10^{-1}$ | | Gharagheizi et al. (2010) | Q | 246 |
| | $6.2\times10^{-2}$ | | Hilal et al. (2008) | Q | |
| | $4.1\times10^{-1}$ | | Modarresi et al. (2007) | Q | 67 |
| | | 6900 | Kühne et al. (2005) | Q | |
| | $9.7\times10^{-2}$ | | Yaffe et al. (2003) | Q | 248, 249 |
| | $9.5\times10^{-2}$ | | English and Carroll (2001) | Q | 230, 274 |
| | $6.7\times10^{-2}$ | | Katritzky et al. (1998) | Q | |
| | $1.8\times10^{-1}$ | | Nirmalakhandan et al. (1997) | Q | |
| | | 8600 | Kühne et al. (2005) | ? | |
| | $9.2\times10^{-2}$ | | Abraham et al. (1990) | ? | |





Table A4.1: Amines (C, H, N) (...continued)

| Substance Formula (Trivial Name) [CAS Registry Number] InChIKey | $H_s^{cp}$ (at $T^{\ominus}$) $\left[\dfrac{\text{mol}}{\text{m}^3\,\text{Pa}}\right]$ | $\dfrac{\text{d}\ln H_s^{cp}}{\text{d}(1/T)}$ [K] | Reference | Type | Note |
|---|---|---|---|---|---|
| N,N-dipropyl-1-propanamine | $2.6\times10^{-2}$ | | Duchowicz et al. (2020) | V | 186 |
| $C_9H_{21}N$ | $2.6\times10^{-2}$ | | HSDB (2015) | V | |
| (tripropylamine) | $2.6\times10^{-2}$ | | Yaws (2003) | X | 237 |
| [102-69-2] | $2.6\times10^{-2}$ | | Hilal et al. (2008) | C | |
| YFTHZRPMJXBUME-UHFFFAOYSA-N | $5.1\times10^{-2}$ | | Duchowicz et al. (2020) | Q | |
| | $3.6\times10^{-2}$ | | Gharagheizi et al. (2012) | Q | |
| | $4.4\times10^{-2}$ | | Gharagheizi et al. (2010) | Q | 246 |
| | $6.7\times10^{-2}$ | | Hilal et al. (2008) | Q | |
| | $1.3\times10^{-2}$ | | Modarresi et al. (2007) | Q | 67 |
| | $2.7\times10^{-2}$ | | Yaffe et al. (2003) | Q | 248, 249 |
| | $2.1\times10^{-2}$ | | Yao et al. (2002) | Q | 229 |
| | $4.2\times10^{-2}$ | | Katritzky et al. (1998) | Q | |
| | $2.6\times10^{-2}$ | | Yaws (1999) | ? | 21 |
| N-methyl-1-butanamine | $1.3\times10^{-1}$ | | Yaws (2003) | X | 237 |
| $C_5H_{13}N$ | $5.7\times10^{-2}$ | | Gharagheizi et al. (2012) | Q | |
| (N-methylbutylamine) | $1.4\times10^{-1}$ | | Gharagheizi et al. (2010) | Q | 246 |
| [110-68-9] | $1.1\times10^{-1}$ | | Hilal et al. (2008) | Q | |
| QCOGKXLOEWLIDC-UHFFFAOYSA-N | | 6600 | Kühne et al. (2005) | Q | |
| | | 5000 | Kühne et al. (2005) | ? | |
| dibutylamine | $1.3\times10^{-1}$ | | Brockbank (2013) | L | |
| $(C_4H_9)_2NH$ | $1.0$ | | Altschuh et al. (1999) | M | |
| [111-92-2] | $1.1\times10^{-1}$ | | Christie and Crisp (1967) | M | |
| JQVDAXLFBXTEQA-UHFFFAOYSA-N | $1.2\times10^{-1}$ | | Mackay et al. (2006d) | V | |
| | $1.2\times10^{-1}$ | | Mackay et al. (1995) | V | |
| | $7.0\times10^{-2}$ | | Yaws (2003) | X | 237 |
| | $4.1\times10^{-1}$ | | Keshavarz et al. (2022) | Q | |
| | $2.2\times10^{-1}$ | | Duchowicz et al. (2020) | Q | 299 |
| | $4.5\times10^{-2}$ | | Gharagheizi et al. (2012) | Q | |
| | $7.6\times10^{-2}$ | | Gharagheizi et al. (2010) | Q | 246 |
| | $2.4\times10^{-1}$ | | Hilal et al. (2008) | Q | |
| | $1.4\times10^{-1}$ | | Modarresi et al. (2007) | Q | 67 |
| | | 7600 | Kühne et al. (2005) | Q | |
| | $9.7\times10^{-2}$ | | Yaffe et al. (2003) | Q | 248, 249 |
| | $6.2\times10^{-2}$ | | Yao et al. (2002) | Q | 229 |
| | $1.2\times10^{-1}$ | | English and Carroll (2001) | Q | 230, 231 |
| | $8.8\times10^{-2}$ | | Katritzky et al. (1998) | Q | |
| | $1.4\times10^{-1}$ | | Nirmalakhandan et al. (1997) | Q | |
| | $1.6\times10^{-1}$ | | Suzuki et al. (1992) | Q | 232 |
| | $1.1\times10^{-1}$ | | Duchowicz et al. (2020) | ? | 185, 21 |
| | | 7400 | Kühne et al. (2005) | ? | |
| | $7.0\times10^{-2}$ | | Yaws (1999) | ? | 21 |
| | $9.7\times10^{-2}$ | | Abraham et al. (1990) | ? | |





Table A4.1: Amines (C, H, N) (... continued)

| Substance Formula (Trivial Name) [CAS Registry Number] InChIKey | $H_s^{cp}$ (at $T^\ominus$) $\left[\dfrac{\mathrm{mol}}{\mathrm{m^3\,Pa}}\right]$ | $\dfrac{\mathrm{d}\ln H_s^{cp}}{\mathrm{d}(1/T)}$ [K] | Reference | Type | Note |
|---|---|---|---|---|---|
| diisobutylamine | $1.8\times10^{-2}$ | | Duchowicz et al. (2020) | V | 186 |
| $C_8H_{19}N$ | $4.3\times10^{-2}$ | | Yaws (2003) | X | 237 |
| [110-96-3] | $3.4\times10^{-2}$ | | Duchowicz et al. (2020) | Q | |
| NJBCRXCAPCODGX-UHFFFAOYSA-N | $6.2\times10^{-2}$ | | Gharagheizi et al. (2010) | Q | 246 |
| | $1.5\times10^{-1}$ | | Modarresi et al. (2007) | Q | 67 |
| | | 7600 | Kühne et al. (2005) | Q | |
| | $1.8\times10^{-2}$ | | Yaffe et al. (2003) | Q | 248, 249 |
| | $1.5\times10^{-2}$ | | Katritzky et al. (1998) | Q | |
| | | 7300 | Kühne et al. (2005) | ? | |
| bis-(1-methylpropyl)-amine | | 7600 | Kühne et al. (2005) | Q | |
| $C_8H_{19}N$ | | 7000 | Kühne et al. (2005) | ? | |
| (di-*sec*-butylamine) | | | | | |
| [626-23-3] | | | | | |
| OBYVIBDTOCAXSN-UHFFFAOYSA-N | | | | | |
| tetraethylenepentamine | $3.3\times10^{14}$ | | HSDB (2015) | Q | 99 |
| $C_8H_{23}N_5$ | | | | | |
| [112-57-2] | | | | | |
| FAGUFWYHJQFNRV-UHFFFAOYSA-N | | | | | |
| N,N-di-2-propenyl-2-propen-1-amine | $3.9\times10^{-2}$ | 7500 | Brockbank (2013) | L | 1 |
| $C_9H_{15}N$ | $3.8\times10^{-2}$ | | Duchowicz et al. (2020) | V | 186 |
| (triallylamine) | $3.8\times10^{-2}$ | | HSDB (2015) | V | |
| [102-70-5] | 2.2 | | Duchowicz et al. (2020) | Q | |
| VPYJNCGUESNPMV-UHFFFAOYSA-N | | | | | |
| trimethylamine | $9.9\times10^{-2}$ | | Burkholder et al. (2019) | L | |
| $(CH_3)_3N$ | $9.9\times10^{-2}$ | | Burkholder et al. (2015) | L | |
| [75-50-3] | $8.4\times10^{-2}$ | 6300 | Leng et al. (2015a) | M | |
| GETQZCLCWQTVFV-UHFFFAOYSA-N | $9.4\times10^{-1}$ | 4200 | Tsuji et al. (1990) | M | 62 |
| | $7.6\times10^{-2}$ | | Amoore and Buttery (1978) | M | |
| | $9.5\times10^{-2}$ | | Christie and Crisp (1967) | M | |
| | $9.8\times10^{-2}$ | | Amoore and Buttery (1978) | V | |
| | $1.0\times10^{-1}$ | | Hayer et al. (2022) | Q | 20 |
| | $9.2\times10^{-2}$ | | Keshavarz et al. (2022) | Q | |
| | 1.3 | | Duchowicz et al. (2020) | Q | |
| | $3.7\times10^{-2}$ | | Hilal et al. (2008) | Q | |
| | $3.3\times10^{-2}$ | | Modarresi et al. (2007) | Q | 67 |
| | $9.7\times10^{-2}$ | | Yaffe et al. (2003) | Q | 248, 249 |
| | $8.4\times10^{-2}$ | | English and Carroll (2001) | Q | 230, 231 |
| | $1.6\times10^{-1}$ | | Katritzky et al. (1998) | Q | |
| | $4.7\times10^{-1}$ | | Nirmalakhandan et al. (1997) | Q | |
| | $6.4\times10^{-2}$ | | Russell et al. (1992) | Q | 279 |
| | $1.2\times10^{-1}$ | | Suzuki et al. (1992) | Q | 232 |
| | $9.5\times10^{-2}$ | | Duchowicz et al. (2020) | ? | 185, 21 |
| | $1.5\times10^{-1}$ | | Mackay et al. (2006d) | ? | |
| | $9.0\times10^{-2}$ | | Abraham et al. (1990) | ? | |



Table A4.1: Amines (C, H, N) (...continued)

| Substance<br>Formula<br>(Trivial Name)<br>[CAS Registry Number]<br>InChIKey | $H_s^{cp}$<br>(at $T^\ominus$)<br>$\left[\dfrac{\text{mol}}{\text{m}^3\,\text{Pa}}\right]$ | $\dfrac{\text{d}\ln H_s^{cp}}{\text{d}(1/T)}$<br><br>[K] | Reference | Type | Note |
|---|---|---|---|---|---|
| triethylamine | $6.6\times10^{-2}$ | | Burkholder et al. (2019) | L | |
| $(C_2H_5)_3N$ | $6.6\times10^{-2}$ | | Burkholder et al. (2015) | L | |
| [121-44-8] | $9.3\times10^{-2}$ | | Brockbank (2013) | L | |
| ZMANZCXQSJIPKH-UHFFFAOYSA-N | $5.0\times10^{-2}$ | 8300 | Leng et al. (2015b) | M | |
| | $6.6\times10^{-2}$ | | Christie and Crisp (1967) | M | |
| | $7.1\times10^{-2}$ | | Mackay et al. (2006d) | V | |
| | $7.1\times10^{-2}$ | | Mackay et al. (1995) | V | |
| | $8.6\times10^{-2}$ | | Yaws (2003) | X | 258 |
| | $8.5\times10^{-2}$ | | Yaws (2003) | X | 237 |
| | $5.0\times10^{-3}$ | | Dupeux et al. (2022) | Q | 259 |
| | $9.2\times10^{-3}$ | | Hayer et al. (2022) | Q | 20 |
| | $2.3\times10^{-1}$ | | Keshavarz et al. (2022) | Q | |
| | $5.1\times10^{-2}$ | | Duchowicz et al. (2020) | Q | 299 |
| | $6.7\times10^{-2}$ | | Li et al. (2014) | Q | 241 |
| | $9.5\times10^{-2}$ | | Gharagheizi et al. (2010) | Q | 246 |
| | $8.6\times10^{-2}$ | | Hilal et al. (2008) | Q | |
| | $2.3\times10^{-2}$ | | Modarresi et al. (2007) | Q | 67 |
| | | 6700 | Kühne et al. (2005) | Q | |
| | $2.1\times10^{-1}$ | | Yaffe et al. (2003) | Q | 248, 272 |
| | $7.8\times10^{-2}$ | | Yao et al. (2002) | Q | 229 |
| | $1.3\times10^{-1}$ | | English and Carroll (2001) | Q | 230, 231 |
| | $8.8\times10^{-2}$ | | Katritzky et al. (1998) | Q | |
| | $3.3\times10^{-1}$ | | Nirmalakhandan et al. (1997) | Q | |
| | $3.1\times10^{-2}$ | | Russell et al. (1992) | Q | 279 |
| | $6.1\times10^{-2}$ | | Suzuki et al. (1992) | Q | 232 |
| | $6.6\times10^{-2}$ | | Duchowicz et al. (2020) | ? | 185, 21 |
| | | 9000 | Kühne et al. (2005) | ? | |
| | $8.5\times10^{-2}$ | | Yaws (1999) | ? | 21 |
| | $9.2\times10^{-2}$ | | Abraham et al. (1990) | ? | |
| tributylamine | $3.7\times10^{-1}$ | 8700 | Brockbank (2013) | L | 1 |
| $C_{12}H_{27}N$ | $4.0\times10^{-1}$ | | Altschuh et al. (1999) | M | |
| [102-82-9] | $6.2\times10^{-2}$ | | Duchowicz et al. (2020) | V | 186 |
| IMFACGCPASFAPR-UHFFFAOYSA-N | $4.0\times10^{-5}$ | | Mackay et al. (2006d) | V | |
| | $4.0\times10^{-5}$ | | Mackay et al. (1995) | V | |
| | $4.9\times10^{-2}$ | | Yaws (2003) | X | 237 |
| | $5.2\times10^{-2}$ | | Duchowicz et al. (2020) | Q | |
| | $2.9\times10^{-2}$ | | Gharagheizi et al. (2010) | Q | 246 |
| | | 8700 | Kühne et al. (2005) | Q | |
| | | 7500 | Kühne et al. (2005) | ? | |
| | $6.1\times10^{-2}$ | | Yaws (1999) | ? | 21 |
| N,N-dimethyl-1-dodecanamine | >4.0 | | Altschuh et al. (1999) | M | |
| $C_{14}H_{31}N$ | $4.6\times10^{-2}$ | | Yaws (2003) | X | 237 |
| [112-18-5] | $2.0\times10^{-3}$ | | HSDB (2015) | Q | 99 |
| YWFWDNVOPHGWMX-UHFFFAOYSA-N | $4.9\times10^{-2}$ | | Gharagheizi et al. (2012) | Q | |
| | $3.5\times10^{-2}$ | | Gharagheizi et al. (2010) | Q | 246 |





Table A4.1: Amines (C, H, N) (... continued)

| Substance Formula (Trivial Name) [CAS Registry Number] InChIKey | $H_s^{cp}$ (at $T^\ominus$) $\left[\dfrac{\text{mol}}{\text{m}^3\,\text{Pa}}\right]$ | $\dfrac{\text{d}\ln H_s^{cp}}{\text{d}(1/T)}$ [K] | Reference | Type | Note |
|---|---|---|---|---|---|
| ethylenediamine | $1.8\times10^3$ | 6700 | Nguyen (2013) | M | 11 |
| $H_2NCH_2CH_2NH_2$ | $5.8\times10^3$ | | Westheimer and Ingraham (1956) | M | |
| [107-15-3] | $1.5\times10^2$ | 9200 | Cabani et al. (1978) | T | |
| PIICEJLVQHRZGT-UHFFFAOYSA-N | $6.7\times10^3$ | | Keshavarz et al. (2022) | Q | |
| | $2.2\times10^3$ | | Duchowicz et al. (2020) | Q | |
| | $5.6\times10^3$ | | Hilal et al. (2008) | Q | |
| | $1.4\times10^4$ | | Modarresi et al. (2007) | Q | 67 |
| | $6.2\times10^2$ | | Yao et al. (2002) | Q | 229 |
| | $5.7\times10^3$ | | Duchowicz et al. (2020) | ? | 185, 21 |
| | $2.3\times10^2$ | | Yaws (1999) | ? | 21, 12 |
| 2-propen-1-amine | 1.0 | 5400 | Leng et al. (2015a) | M | |
| $C_3H_7N$ | $5.4\times10^{-1}$ | | Duchowicz et al. (2020) | V | 186 |
| (allylamine) | $5.4\times10^{-1}$ | | HSDB (2015) | V | |
| [107-11-9] | $5.4\times10^{-1}$ | | Hilal et al. (2008) | C | |
| VVJKKWFAADXIJK-UHFFFAOYSA-N | 7.7 | | Duchowicz et al. (2020) | Q | |
| | 2.4 | | Hilal et al. (2008) | Q | |
| | 4.4 | | Modarresi et al. (2007) | Q | 67 |
| di-2-propenylamine | $3.3\times10^{-1}$ | | Duchowicz et al. (2020) | V | 186 |
| $C_6H_{11}N$ | $3.3\times10^{-1}$ | | HSDB (2015) | V | |
| (diallylamine) | 2.6 | | Duchowicz et al. (2020) | Q | |
| [124-02-7] | | 7200 | Kühne et al. (2005) | Q | |
| DYUWTXWIYMHBQS-UHFFFAOYSA-N | | 8000 | Kühne et al. (2005) | ? | |
| hexamethyleneimine | 1.6 | 8200 | Cabani et al. (1971a) | T | |
| $(CH_2)_6NH$ | 1.0 | | Keshavarz et al. (2022) | Q | |
| [111-49-9] | $1.3\times10^1$ | | Duchowicz et al. (2020) | Q | 184 |
| ZSIQJIWKELUFRJ-UHFFFAOYSA-N | 6.4 | | Hilal et al. (2008) | Q | |
| | $3.5\times10^{-1}$ | | Modarresi et al. (2007) | Q | 67 |
| | 1.2 | | Suzuki et al. (1992) | Q | 232 |
| | $4.3\times10^{-1}$ | | Meylan and Howard (1991) | Q | |
| | 1.6 | | Duchowicz et al. (2020) | ? | 185, 21 |
| cyclohexanamine | 2.2 | 7500 | Brockbank (2013) | L | 1 |
| $C_6H_{13}N$ | 2.4 | | Altschuh et al. (1999) | M | |
| (cyclohexylamine) | 2.2 | 7800 | Bernauer et al. (2006) | V | 1 |
| [108-91-8] | $9.4\times10^{-1}$ | | Amoore and Buttery (1978) | V | |
| PAFZNILMFXTMIY-UHFFFAOYSA-N | 3.3 | | Keshavarz et al. (2022) | Q | |
| | 7.3 | | Duchowicz et al. (2020) | Q | 299 |
| | $6.7\times10^{-1}$ | | Hilal et al. (2008) | Q | |
| | 4.2 | | Modarresi et al. (2007) | Q | 67 |
| | $9.5\times10^{-1}$ | | English and Carroll (2001) | Q | 230, 231 |
| | 1.2 | | Nirmalakhandan et al. (1997) | Q | |
| | 2.4 | | Duchowicz et al. (2020) | ? | 185, 21 |
| | $9.5\times10^{-1}$ | | Abraham et al. (1990) | ? | |





Table A4.1: Amines (C, H, N) (... continued)

| Substance Formula (Trivial Name) [CAS Registry Number] InChIKey | $H_s^{cp}$ (at $T^{\ominus}$) $\left[\dfrac{\text{mol}}{\text{m}^3\,\text{Pa}}\right]$ | $\dfrac{\text{d}\ln H_s^{cp}}{\text{d}(1/T)}$ [K] | Reference | Type | Note |
|---|---|---|---|---|---|
| 3-methylcyclohexylamine $C_7H_{15}N$ [6850-35-7] JYDYHSHPBDZRPU-UHFFFAOYSA-N | 1.1 | | Hilal et al. (2008) | Q | |
| N-ethylcyclohexanamine $C_8H_{17}N$ (N-ethylcyclohexylamine) [5459-93-8] AGVKXDPPPSLISR-UHFFFAOYSA-N | | 7200 6500 | Kühne et al. (2005) Kühne et al. (2005) | Q ? | |
| N,N-dimethylcyclohexylamine $C_8H_{17}N$ [98-94-2] SVYKKECYCPFKGB-UHFFFAOYSA-N | $4.2\times10^{-1}$ $4.1\times10^{-1}$ 1.4 $5.1\times10^{-1}$ $8.5\times10^{-2}$ $4.2\times10^{-1}$ | 7000 8500 | Altschuh et al. (1999) Keshavarz et al. (2022) Duchowicz et al. (2020) Hilal et al. (2008) Modarresi et al. (2007) Kühne et al. (2005) Duchowicz et al. (2020) Kühne et al. (2005) | M Q Q Q Q Q ? ? | 299 67 185, 21 |
| hexamethylenetetramine $C_6H_{12}N_4$ [100-97-0] VKYKSIONXSXAKP-UHFFFAOYSA-N | $6.2\times10^3$ $6.1\times10^{-5}$ $5.8\times10^5$ $9.2\times10^2$ $5.4\times10^7$ $1.3\times10^4$ $5.8\times10^3$ | | HSDB (2015) Zhang et al. (2010) Zhang et al. (2010) Zhang et al. (2010) Zhang et al. (2010) Hilal et al. (2008) Modarresi et al. (2007) | V Q Q Q Q Q Q | 287, 288 287, 289 287, 290 287, 291 67 |
| 1-decanamine $C_{10}H_{23}N$ [2016-57-1] MHZGKXUYDGKKIU-UHFFFAOYSA-N | $2.6\times10^{-1}$ $1.5\times10^{-1}$ 2.4 | | Duchowicz et al. (2020) Yaws et al. (2001) Duchowicz et al. (2020) | V X Q | 186 350 |
| N-cyclohexylcyclohexanamine $C_{12}H_{23}N$ (dicyclohexylamine) [101-83-7] XBPCUCUWBYBCDP-UHFFFAOYSA-N | $1.8\times10^{-1}$ | | HSDB (2015) | Q | 99 |
| 1-dodecanamine $C_{12}H_{27}N$ [124-22-1] JRBPAEWTRLWTQC-UHFFFAOYSA-N | $3.7\times10^{-2}$ | | HSDB (2015) | Q | 99 |
| 1-octadecanamine $C_{18}H_{39}N$ [124-30-1] REYJJPSVUYRZGE-UHFFFAOYSA-N | $1.0\times10^{-2}$ | | HSDB (2015) | Q | 99 |



Table A4.1: Amines (C, H, N) (. . . continued)

| Substance<br>Formula<br>(Trivial Name)<br>[CAS Registry Number]<br>InChIKey | $H_s^{cp}$<br>(at $T^\ominus$)<br>$\left[\dfrac{\mathrm{mol}}{\mathrm{m^3\,Pa}}\right]$ | $\dfrac{\mathrm{d}\ln H_s^{cp}}{\mathrm{d}(1/T)}$<br><br>[K] | Reference | Type | Note |
|---|---|---|---|---|---|
| N,N-dioctyl-1-octanamine<br>$C_{24}H_{51}N$<br>(tri-N-octylamine)<br>[1116-76-3]<br>XTAZYLNFDRKIHJ-UHFFFAOYSA-N | $7.0\times10^{-4}$ | | HSDB (2015) | Q | 99 |
| aminobenzene<br>$C_6H_7N$<br>(aniline)<br>[62-53-3]<br>PAYRUJLWNCNPSJ-UHFFFAOYSA-N | 4.9 | 6800 | Brockbank (2013) | L | 1 |
| | 4.4 | | Chao et al. (2017) | M | |
| | 5.2 | | Altschuh et al. (1999) | M | |
| | 1.2 | | Heal et al. (1995) | M | 373 |
| | 5.0 | | Jayasinghe et al. (1992) | M | |
| | 1.1 | | Dallos et al. (1983) | M | 553 |
| | 4.4 | | Chao et al. (2017) | V | |
| | 4.6 | 6500 | Bernauer et al. (2006) | V | 1 |
| | 6.0 | | Mackay et al. (2006d) | V | |
| | 6.0 | | Schüürmann (2000) | V | |
| | $7.1\times10^{-5}$ | | Lide and Frederikse (1995) | V | |
| | 6.0 | | Mackay et al. (1995) | V | |
| | 5.5 | | Hwang et al. (1992) | V | |
| | 3.4 | | Yoshida et al. (1983) | V | |
| | 5.8 | | Yaws (2003) | X | 258 |
| | $7.1\times10^{-5}$ | | Howard (1989) | X | 364 |
| | 5.7 | | Yaws (2003) | X | 237 |
| | $8.2\times10^{-2}$ | | Howard (1989) | X | 412 |
| | 2.4 | | Dupeux et al. (2022) | Q | 259 |
| | 3.3 | | Keshavarz et al. (2022) | Q | |
| | 5.1 | | Duchowicz et al. (2020) | Q | 299 |
| | $1.8\times10^1$ | | Gharagheizi et al. (2012) | Q | |
| | 5.5 | | Gharagheizi et al. (2010) | Q | 246 |
| | 5.1 | | Hilal et al. (2008) | Q | |
| | 4.2 | | Modarresi et al. (2007) | Q | 67 |
| | | 6200 | Kühne et al. (2005) | Q | |
| | 4.8 | | Yaffe et al. (2003) | Q | 248, 249 |
| | 2.4 | | Yao et al. (2002) | Q | 229 |
| | 3.0 | | Katritzky et al. (1998) | Q | |
| | 4.9 | | Duchowicz et al. (2020) | ? | 185, 21 |
| | $8.2\times10^{-2}$ | | Mackay et al. (2006d) | ? | |
| | | 7100 | Kühne et al. (2005) | ? | |
| | 5.8 | | Yaws (1999) | ? | 21 |
| | 4.3 | | Abraham et al. (1990) | ? | |
| benzylamine<br>$C_7H_9N$<br>[100-46-9]<br>WGQKYBSKWIADBV-UHFFFAOYSA-N | $7.1\times10^{-1}$ | | Yao et al. (2002) | Q | 229 |
| | $3.4\times10^{-1}$ | | Yaws (1999) | ? | 21 |



Table A4.1: Amines (C, H, N) (... continued)

| Substance Formula (Trivial Name) [CAS Registry Number] InChIKey | $H_s^{cp}$ (at $T^\ominus$) $\left[\dfrac{\mathrm{mol}}{\mathrm{m^3\,Pa}}\right]$ | $\dfrac{\mathrm{d}\ln H_s^{cp}}{\mathrm{d}(1/T)}$ [K] | Reference | Type | Note |
|---|---|---|---|---|---|
| 2-methylbenzenamine | 4.8 | 7100 | Brockbank (2013) | L | 1 |
| $C_7H_9N$ | 3.0 | | Chao et al. (2017) | M | |
| (2-methylaniline; $o$-toluidine) | 5.0 | | Altschuh et al. (1999) | M | |
| [95-53-4] | 4.0 | | Chao et al. (2017) | V | |
| RNVCVTLRINQCPJ-UHFFFAOYSA-N | $1.1\times10^1$ | | Mackay et al. (2006d) | V | |
| | 4.1 | | Schüürmann (2000) | V | |
| | $1.1\times10^1$ | | Mackay et al. (1995) | V | |
| | $1.1\times10^1$ | | Mackay et al. (1995) | V | |
| | 3.4 | | Yoshida et al. (1983) | V | |
| | 4.6 | | Abraham et al. (1994a) | R | |
| | 4.1 | | Yaws (2003) | X | 237 |
| | $3.0\times10^{-1}$ | | Keshavarz et al. (2022) | Q | |
| | 2.7 | | Duchowicz et al. (2020) | Q | 299 |
| | 3.3 | | Gharagheizi et al. (2012) | Q | |
| | 4.4 | | Gharagheizi et al. (2010) | Q | 246 |
| | 3.1 | | Hilal et al. (2008) | Q | |
| | 1.0 | | Modarresi et al. (2007) | Q | 67 |
| | 1.6 | | Yao et al. (2002) | Q | 229 |
| | 4.0 | | English and Carroll (2001) | Q | 230, 231 |
| | 4.4 | | Katritzky et al. (1998) | Q | |
| | 2.0 | | Nirmalakhandan et al. (1997) | Q | |
| | 5.0 | | Duchowicz et al. (2020) | ? | 185, 21 |
| | 4.1 | | Yaws (1999) | ? | 21 |
| 3-methylbenzenamine | 5.6 | 7200 | Brockbank (2013) | L | 1 |
| $C_7H_9N$ | 3.2 | | Chao et al. (2017) | M | |
| (3-methylaniline; $m$-toluidine) | 5.9 | | Altschuh et al. (1999) | M | |
| [108-44-1] | 3.9 | | Chao et al. (2017) | V | |
| JJYPMNFTHPTTDI-UHFFFAOYSA-N | 3.9 | | Mackay et al. (2006d) | V | |
| | 3.9 | | Mackay et al. (1995) | V | |
| | 5.1 | | Yaws (2003) | X | 237, 12 |
| | 4.5 | | Keshavarz et al. (2022) | Q | |
| | 2.7 | | Duchowicz et al. (2020) | Q | 299 |
| | 4.7 | | Gharagheizi et al. (2012) | Q | |
| | 4.4 | | Gharagheizi et al. (2010) | Q | 246 |
| | 4.8 | | Hilal et al. (2008) | Q | |
| | 3.9 | | Modarresi et al. (2007) | Q | 67 |
| | 1.5 | | Yao et al. (2002) | Q | 229 |
| | 6.4 | | Katritzky et al. (1998) | Q | |
| | 5.9 | | Duchowicz et al. (2020) | ? | 185, 21 |
| | 3.5 | | Yaws (1999) | ? | 21, 12 |





Table A4.1: Amines (C, H, N) (...continued)

| Substance Formula (Trivial Name) [CAS Registry Number] InChIKey | $H_s^{cp}$ (at $T^\ominus$) $\left[\dfrac{mol}{m^3\,Pa}\right]$ | $\dfrac{d\ln H_s^{cp}}{d(1/T)}$ [K] | Reference | Type | Note |
|---|---|---|---|---|---|
| 4-methylbenzenamine | 2.9 | 4400 | Brockbank (2013) | L | 1 |
| $C_7H_9N$ | 4.4 | | Chao et al. (2017) | M | |
| (4-methylaniline; $p$-toluidine) | $1.3\times10^1$ | | Altschuh et al. (1999) | M | |
| [106-49-0] | 4.4 | | Jayasinghe et al. (1992) | M | |
| RZXMPPFPUUCRFN-UHFFFAOYSA-N | 1.5 | | Mackay et al. (2006d) | V | |
| | 1.5 | | Mackay et al. (1995) | V | |
| | 1.6 | | Yoshida et al. (1983) | V | |
| | 5.0 | | Abraham et al. (1994a) | R | |
| | 4.5 | | Keshavarz et al. (2022) | Q | |
| | 2.7 | | Duchowicz et al. (2020) | Q | |
| | 3.9 | | Gharagheizi et al. (2012) | Q | |
| | 5.3 | | Hilal et al. (2008) | Q | |
| | 5.0 | | Modarresi et al. (2007) | Q | 67 |
| | 4.0 | | English and Carroll (2001) | Q | 230, 231 |
| | 6.1 | | Katritzky et al. (1998) | Q | |
| | 2.0 | | Nirmalakhandan et al. (1997) | Q | |
| | 4.9 | | Duchowicz et al. (2020) | ? | 185, 21 |
| 2-ethylaniline | 2.6 | 7300 | Brockbank (2013) | L | 1, 554 |
| $C_8H_{11}N$ | 2.7 | | HSDB (2015) | Q | 99 |
| ($o$-ethylaniline) | | 7200 | Kühne et al. (2005) | Q | |
| [578-54-1] | | 7500 | Kühne et al. (2005) | ? | |
| MLPVBIWIRCKMJV-UHFFFAOYSA-N | | | | | |
| 4-ethylaniline | 3.1 | | Mackay et al. (2006d) | V | |
| $C_8H_{11}N$ | 3.1 | | Mackay et al. (1995) | V | |
| ($p$-ethylaniline) | | 6900 | Kühne et al. (2005) | Q | |
| [589-16-2] | | 8100 | Kühne et al. (2005) | ? | |
| HRXZRAXKKNUKRF-UHFFFAOYSA-N | | | | | |
| 2,4-dimethylbenzenamine | 2.8 | 7700 | Brockbank (2013) | L | 1 |
| $C_8H_{11}N$ | 2.4 | | Mackay et al. (2006d) | V | |
| (2,4-dimethylaniline; 2,4-xylidine) | $1.4\times10^{-1}$ | | Schüürmann (2000) | V | |
| [95-68-1] | 2.4 | | Mackay et al. (1995) | V | |
| CZZZABOKJQXEBO-UHFFFAOYSA-N | 3.9 | | HSDB (2015) | Q | 99 |
| | | 7200 | Kühne et al. (2005) | Q | |
| | | 7400 | Kühne et al. (2005) | ? | |
| 3,4-dimethylbenzenamine | 5.3 | | Jayasinghe et al. (1992) | M | |
| $C_8H_{11}N$ | 6.0 | | Keshavarz et al. (2022) | Q | |
| (3,4-dimethylaniline; 3,4-xylidine) | 1.4 | | Duchowicz et al. (2020) | Q | |
| [95-64-7] | 6.7 | | Hilal et al. (2008) | Q | |
| DOLQYFPDPKPQSS-UHFFFAOYSA-N | 1.6 | | Modarresi et al. (2007) | Q | 67 |
| | 9.0 | | Katritzky et al. (1998) | Q | |
| | 5.3 | | Duchowicz et al. (2020) | ? | 185, 21 |



Table A4.1: Amines (C, H, N) (...continued)

| Substance Formula (Trivial Name) [CAS Registry Number] InChIKey | $H_s^{cp}$ (at $T^{\ominus}$) $\left[\dfrac{\text{mol}}{\text{m}^3\,\text{Pa}}\right]$ | $\dfrac{\mathrm{d}\ln H_s^{cp}}{\mathrm{d}(1/T)}$ [K] | Reference | Type | Note |
|---|---|---|---|---|---|
| 2,5-dimethylbenzenamine | 2.3 | | Duchowicz et al. (2020) | V | 186 |
| $C_8H_{11}N$ | 1.4 | | Duchowicz et al. (2020) | Q | |
| (2,5-dimethylaniline; 2,5-xylidine) | 3.9 | | HSDB (2015) | Q | 99 |
| [95-78-3] | $7.7\times10^{-1}$ | | Modarresi et al. (2007) | Q | 67 |
| VOWZNBNDMFLQGM-UHFFFAOYSA-N | | 7200 | Kühne et al. (2005) | Q | |
| | | 7700 | Kühne et al. (2005) | ? | |
| 2,6-dimethylbenzenamine | 3.9 | | Duchowicz et al. (2020) | V | 186 |
| $C_8H_{11}N$ | 3.9 | | HSDB (2015) | V | |
| (2,6-dimethylaniline; 2,6-xylidine) | $5.8\times10^{-2}$ | | Mackay et al. (2006d) | V | |
| [87-62-7] | $5.8\times10^{-2}$ | | Mackay et al. (1995) | V | |
| UFFBMTHBGFGIHF-UHFFFAOYSA-N | 2.7 | | Abraham et al. (1994a) | R | |
| | $2.9\times10^{-1}$ | | Duchowicz et al. (2020) | Q | |
| | 3.3 | | Hilal et al. (2008) | Q | |
| | $1.7\times10^{1}$ | | Modarresi et al. (2007) | Q | 67 |
| | | 7500 | Kühne et al. (2005) | Q | |
| | 2.1 | | English and Carroll (2001) | Q | 230, 231 |
| | 4.2 | | Katritzky et al. (1998) | Q | |
| | 1.4 | | Nirmalakhandan et al. (1997) | Q | |
| | | 7600 | Kühne et al. (2005) | ? | |
| 2,4,5-trimethylbenzenamine | 3.9 | | Jayasinghe et al. (1992) | M | |
| $C_9H_{13}N$ | $5.5\times10^{-1}$ | | Keshavarz et al. (2022) | Q | |
| (2,4,5-trimethylaniline) | $7.3\times10^{-1}$ | | Duchowicz et al. (2020) | Q | 184 |
| [137-17-7] | 6.0 | | Hilal et al. (2008) | Q | |
| BMIPMKQAAJKBKP-UHFFFAOYSA-N | $3.5\times10^{-1}$ | | Modarresi et al. (2007) | Q | 67 |
| | 4.0 | | Duchowicz et al. (2020) | ? | 185, 21 |
| 2-(1-methylethyl)-benzenamine | | 7500 | Kühne et al. (2005) | Q | |
| $C_9H_{13}N$ | | 6400 | Kühne et al. (2005) | ? | |
| (2-isopropylaniline) | | | | | |
| [643-28-7] | | | | | |
| YKOLZVXSPGIIBJ-UHFFFAOYSA-N | | | | | |
| 2,6-diethylbenzenamine | 8.8 | | Duchowicz et al. (2020) | V | 186 |
| $C_{10}H_{15}N$ | 9.0 | | HSDB (2015) | V | |
| (2,6-diethylaniline) | $3.0\times10^{-1}$ | | Duchowicz et al. (2020) | Q | |
| [579-66-8] | $9.0\times10^{-1}$ | | Hilal et al. (2008) | Q | |
| FOYHNROGBXVLLX-UHFFFAOYSA-N | 6.8 | | Modarresi et al. (2007) | Q | 67 |
| | 8.8 | | Yaffe et al. (2003) | Q | 248, 249 |
| | 5.0 | | Katritzky et al. (1998) | Q | |
| | 8.8 | | Yaws (1999) | ? | 21, 402 |
| 1,2-benzenediamine | $1.4\times10^{3}$ | | Duchowicz et al. (2020) | V | 186 |
| $C_6H_8N_2$ | $1.4\times10^{3}$ | | HSDB (2015) | V | |
| (o-phenylenediamine) | $7.6\times10^{1}$ | | Schüürmann (2000) | V | |
| [95-54-5] | $1.0\times10^{2}$ | | Yaws (2003) | X | 237, 555 |
| GEYOCULIXLDCMW-UHFFFAOYSA-N | $2.9\times10^{3}$ | | Duchowicz et al. (2020) | Q | |
| | $1.2\times10^{2}$ | | Gharagheizi et al. (2012) | Q | |
| | $1.0\times10^{2}$ | | Gharagheizi et al. (2010) | Q | 246 |





Table A4.1: Amines (C, H, N) (...continued)

| Substance Formula (Trivial Name) [CAS Registry Number] InChIKey | $H_s^{cp}$ (at $T^\ominus$) $\left[\dfrac{\text{mol}}{\text{m}^3\,\text{Pa}}\right]$ | $\dfrac{\mathrm{d}\ln H_s^{cp}}{\mathrm{d}(1/T)}$ [K] | Reference | Type | Note |
|---|---|---|---|---|---|
| | $1.2\times10^3$ | | Hilal et al. (2008) | Q | |
| 1,3-benzenediamine | $7.9\times10^3$ | | Duchowicz et al. (2020) | V | 186 |
| $C_6H_8N_2$ | $7.6\times10^3$ | | HSDB (2015) | V | |
| ($m$-phenylenediamine) | $1.3\times10^4$ | | Schüürmann (2000) | V | |
| [108-45-2] | $6.9\times10^3$ | | Yaws (2003) | X | 237 |
| WZCQRUWWHSTZEM-UHFFFAOYSA-N | $2.8\times10^3$ | | Duchowicz et al. (2020) | Q | |
| | $2.8\times10^2$ | | Gharagheizi et al. (2012) | Q | |
| | $2.2\times10^3$ | | Gharagheizi et al. (2010) | Q | 246 |
| | $1.1\times10^5$ | | Hilal et al. (2008) | Q | |
| | $3.7\times10^4$ | | Modarresi et al. (2007) | Q | 67 |
| 1,4-benzenediamine | $7.1\times10^2$ | | Yaws (2003) | X | 237, 79 |
| $C_6H_8N_2$ | $1.5\times10^4$ | | HSDB (2015) | Q | 99 |
| ($p$-phenylenediamine) | $1.7\times10^2$ | | Gharagheizi et al. (2012) | Q | |
| [106-50-3] | $2.2\times10^3$ | | Gharagheizi et al. (2010) | Q | 246 |
| CBCKQZAAMUWICA-UHFFFAOYSA-N | | | | | |
| 2-methyl-1,3-benzenediamine | $1.3\times10^4$ | | HSDB (2015) | Q | 99 |
| $C_7H_{10}N_2$ | | | | | |
| [823-40-5] | | | | | |
| RLYCRLGLCUXUPO-UHFFFAOYSA-N | | | | | |
| 2-methyl-1,4-benzenediamine | $1.3\times10^4$ | | HSDB (2015) | Q | 99 |
| $C_7H_{10}N_2$ | | | | | |
| [95-70-5] | | | | | |
| OBCSAIDCZQSFQH-UHFFFAOYSA-N | | | | | |
| 3-methyl-1,2-benzenediamine | $1.0\times10^4$ | | HSDB (2015) | Q | 99 |
| $C_7H_{10}N_2$ | | | | | |
| (2,3-diaminotoluene) | | | | | |
| [2687-25-4] | | | | | |
| AXNUJYHFQHQZBE-UHFFFAOYSA-N | | | | | |
| 4-methyl-1,3-benzenediamine | $1.0\times10^4$ | | HSDB (2015) | Q | 99 |
| $C_7H_{10}N_2$ | | | | | |
| (toluene-2,4-diamine) | | | | | |
| [95-80-7] | | | | | |
| VOZKAJLKRJDJLL-UHFFFAOYSA-N | | | | | |
| 3,5-diaminotoluene | $1.3\times10^4$ | | HSDB (2015) | Q | 545 |
| $C_7H_{10}N_2$ | | | | | |
| [108-71-4] | | | | | |
| LVNDUJYMLJDECN-UHFFFAOYSA-N | | | | | |
| phenylhydrazine | $2.2\times10^3$ | | Duchowicz et al. (2020) | V | 186 |
| $C_6H_8N_2$ | $3.4\times10^2$ | | HSDB (2015) | V | |
| [100-63-0] | 9.6 | | Duchowicz et al. (2020) | Q | |
| HKOOXMFOFWEVGF-UHFFFAOYSA-N | $6.9\times10^2$ | | Hilal et al. (2008) | Q | |
| | $8.3\times10^3$ | | Modarresi et al. (2007) | Q | 67 |



Table A4.1: Amines (C, H, N) (... continued)

| Substance Formula (Trivial Name) [CAS Registry Number] InChIKey | $H_s^{cp}$ (at $T^{\ominus}$) $\left[\dfrac{\mathrm{mol}}{\mathrm{m^3\,Pa}}\right]$ | $\dfrac{\mathrm{d}\ln H_s^{cp}}{\mathrm{d}(1/T)}$ [K] | Reference | Type | Note |
|---|---|---|---|---|---|
| (methylamino)-benzene | $8.7\times10^{-1}$ | | HSDB (2015) | V | |
| $C_7H_9N$ | $8.7\times10^{-1}$ | | Schüürmann (2000) | V | |
| (N-methylaniline) | 1.1 | | Abraham et al. (1994a) | R | |
| [100-61-8] | $3.0\times10^{-1}$ | | Keshavarz et al. (2022) | Q | |
| AFBPFSWMIHJQDM-UHFFFAOYSA-N | 3.7 | | Duchowicz et al. (2020) | Q | 299 |
| | 1.5 | | Hilal et al. (2008) | Q | |
| | $5.7\times10^{-1}$ | | Modarresi et al. (2007) | Q | 67 |
| | $6.1\times10^{-1}$ | | Yaffe et al. (2003) | Q | 248, 272 |
| | 2.0 | | Yao et al. (2002) | Q | 229 |
| | $7.3\times10^{-1}$ | | English and Carroll (2001) | Q | 230, 231 |
| | 4.5 | | Katritzky et al. (1998) | Q | |
| | 2.7 | | Nirmalakhandan et al. (1997) | Q | |
| | 1.1 | | Duchowicz et al. (2020) | ? | 185, 21 |
| | $8.7\times10^{-1}$ | | Yaws (1999) | ? | 21 |
| (ethylamino)-benzene | $8.1\times10^{-1}$ | 7900 | Brockbank (2013) | L | 1, 556 |
| $C_8H_{11}N$ | 1.0 | | Altschuh et al. (1999) | M | |
| (N-ethylaniline) | $4.1\times10^{-1}$ | | Keshavarz et al. (2022) | Q | |
| [103-69-5] | 1.3 | | Duchowicz et al. (2020) | Q | 184 |
| OJGMBLNIHDZDGS-UHFFFAOYSA-N | $6.2\times10^{-1}$ | | HSDB (2015) | Q | 99 |
| | $7.0\times10^{-1}$ | | Hilal et al. (2008) | Q | |
| | $9.5\times10^{-1}$ | | Modarresi et al. (2007) | Q | 67 |
| | | 7100 | Kühne et al. (2005) | Q | |
| | $6.1\times10^{-1}$ | | Yaffe et al. (2003) | Q | 248, 249 |
| | 3.1 | | Katritzky et al. (1998) | Q | |
| | 1.0 | | Duchowicz et al. (2020) | ? | 185, 21 |
| | | 7600 | Kühne et al. (2005) | ? | |
| (dimethylamino)-benzene | $1.7\times10^{-1}$ | | HSDB (2015) | V | |
| $C_8H_{11}N$ | $8.5\times10^{-2}$ | | Mackay et al. (2006d) | V | |
| (N,N-dimethylaniline) | $8.5\times10^{-2}$ | | Mackay et al. (1995) | V | |
| [121-69-7] | $1.3\times10^{-1}$ | | Meylan and Howard (1991) | V | |
| JLTDJTHDQAWBAV-UHFFFAOYSA-N | $1.6\times10^{-1}$ | | Yoshida et al. (1983) | V | |
| | $9.9\times10^{-2}$ | | Hilal et al. (2008) | Q | |
| | $1.0\times10^{-1}$ | | Modarresi et al. (2007) | Q | 67 |
| | | 6900 | Kühne et al. (2005) | Q | |
| | $1.4\times10^{-1}$ | | Yaffe et al. (2003) | Q | 248, 249 |
| | 2.5 | | Katritzky et al. (1998) | Q | |
| | 2.4 | | Nirmalakhandan et al. (1997) | Q | |
| | 1.1 | | Meylan and Howard (1991) | Q | |
| | | 6300 | Kühne et al. (2005) | ? | |
| | $9.7\times10^{-2}$ | | Yaws (1999) | ? | 21 |
| | $1.4\times10^{-1}$ | | Abraham et al. (1990) | ? | |





Table A4.1: Amines (C, H, N) (. . . continued)

| Substance Formula (Trivial Name) [CAS Registry Number] InChIKey | $H_s^{cp}$ (at $T^{\ominus}$) $\left[\dfrac{\text{mol}}{\text{m}^3\,\text{Pa}}\right]$ | $\dfrac{\text{d}\ln H_s^{cp}}{\text{d}(1/T)}$ [K] | Reference | Type | Note |
|---|---|---|---|---|---|
| benzeneethanamine $C_8H_{11}N$ (2-phenylethylamine) [64-04-0] BHHGXPLMPWCGHP-UHFFFAOYSA-N | $1.2\times10^1$ | | HSDB (2015) | Q | 99 |
| 2,3-dimethylbenzenamine $C_8H_{11}N$ (2,3-xylidine) [87-59-2] VVAKEQGKZNKUSU-UHFFFAOYSA-N | 3.9 | | HSDB (2015) | Q | 99 |
| 3,5-dimethylbenzenamine $C_8H_{11}N$ [108-69-0] MKARNSWMMBGSHX-UHFFFAOYSA-N | 3.9 | | HSDB (2015) | Q | 99 |
| dimethylaniline $C_8H_{11}N$ (xylidine) [1300-73-8] CDULGHZNHURECF-UHFFFAOYSA-N | 3.9 | | HSDB (2015) | Q | 99 |
| phenelzine $C_8H_{12}N_2$ [51-71-8] RMUCZJUITONUFY-UHFFFAOYSA-N | $2.9\times10^3$ | | HSDB (2015) | Q | 99 |
| N,N-dimethyl-1,4-benzenediamine $C_8H_{12}N_2$ [99-98-9] BZORFPDSXLZWJF-UHFFFAOYSA-N | $3.3\times10^2$ | | HSDB (2015) | Q | 99 |
| 2,4,6-trimethylbenzenamine $C_9H_{13}N$ (2,4,6-trimethylaniline) [88-05-1] KWVPRPSXBZNOHS-UHFFFAOYSA-N | 3.7 | | HSDB (2015) | Q | 99 |
| N-ethyl-3-methylbenzenamine $C_9H_{13}N$ [102-27-2] GUYMMHOQXYZMJQ-UHFFFAOYSA-N | 1.6 | | HSDB (2015) | Q | 99 |
| N-(1-methylethyl)benzenamine $C_9H_{13}N$ [768-52-5] FRCFWPVMFJMNDP-UHFFFAOYSA-N | 1.3 | | HSDB (2015) | Q | 99 |



Table A4.1: Amines (C, H, N) (...continued)

| Substance Formula (Trivial Name) [CAS Registry Number] InChIKey | $H_s^{cp}$ (at $T^{\ominus}$) $\left[\dfrac{\text{mol}}{\text{m}^3\,\text{Pa}}\right]$ | $\dfrac{\mathrm{d}\ln H_s^{cp}}{\mathrm{d}(1/T)}$ [K] | Reference | Type | Note |
|---|---|---|---|---|---|
| 2-ethyl-6-methylbenzenamine | 3.2 | | Zhang et al. (2010) | Q | 287, 288 |
| $C_9H_{13}N$ | 2.1 | | Zhang et al. (2010) | Q | 287, 289 |
| [24549-06-2] | 1.0 | | Zhang et al. (2010) | Q | 287, 290 |
| JJVKJJNCIILLRP-UHFFFAOYSA-N | $8.4\times10^{-1}$ | | Zhang et al. (2010) | Q | 287, 291 |
| N,N-dimethylbenzylamine | | 7700 | Kühne et al. (2005) | Q | |
| $C_9H_{13}N$ | | 7700 | Kühne et al. (2005) | ? | |
| [103-83-3] | | | | | |
| XXBDWLFCJWSEKW-UHFFFAOYSA-N | | | | | |
| N,N,4-trimethylbenzenamine | $2.0\times10^{-1}$ | | Duchowicz et al. (2020) | V | 186 |
| $C_9H_{13}N$ | $1.1\times10^{-1}$ | | Duchowicz et al. (2020) | Q | |
| [99-97-8] | $1.4\times10^{-1}$ | | Hilal et al. (2008) | Q | |
| GYVGXEWAOAAJEU-UHFFFAOYSA-N | $4.0\times10^{-2}$ | | Modarresi et al. (2007) | Q | 67 |
| N,N'-di-*tert*-butylethylenediamine | $3.6\times10^{2}$ | | Zhang et al. (2010) | Q | 287, 288 |
| $C_{10}H_{24}N_2$ | 2.3 | | Zhang et al. (2010) | Q | 287, 289 |
| [4062-60-6] | $9.9\times10^{-1}$ | | Zhang et al. (2010) | Q | 287, 290 |
| KGHYGBGIWLNFAV-UHFFFAOYSA-N | $1.2\times10^{1}$ | | Zhang et al. (2010) | Q | 287, 291 |
| (diethylamino)-benzene | $6.7\times10^{-2}$ | 6700 | Brockbank (2013) | L | 1 |
| $C_{10}H_{15}N$ | $5.1\times10^{-2}$ | | Duchowicz et al. (2020) | V | 186 |
| (N,N-diethylaniline) | $5.2\times10^{-2}$ | | HSDB (2015) | V | |
| [91-66-7] | $4.6\times10^{-1}$ | | Mackay et al. (2006d) | V | |
| GGSUCNLOZRCGPQ-UHFFFAOYSA-N | $4.6\times10^{-1}$ | | Mackay et al. (1995) | V | |
| | $2.4\times10^{-2}$ | | Duchowicz et al. (2020) | Q | |
| | $9.9\times10^{-2}$ | | Hilal et al. (2008) | Q | |
| | $6.0\times10^{-2}$ | | Modarresi et al. (2007) | Q | 67 |
| | | 7600 | Kühne et al. (2005) | Q | |
| | 2.4 | | Katritzky et al. (1998) | Q | |
| | | 5800 | Kühne et al. (2005) | ? | |
| | 5.2 | | Yaws (1999) | ? | 21, 557 |
| 1-naphthylamine | $1.6\times10^{2}$ | | Altschuh et al. (1999) | M | |
| $C_{10}H_9N$ | $2.1\times10^{1}$ | | HSDB (2015) | V | |
| [134-32-7] | $8.8\times10^{1}$ | | Abraham et al. (1994a) | R | |
| RUFPHBVGCFYCNW-UHFFFAOYSA-N | $6.2\times10^{2}$ | | Keshavarz et al. (2022) | Q | |
| | $2.0\times10^{1}$ | | Duchowicz et al. (2020) | Q | 184 |
| | $3.0\times10^{1}$ | | Hilal et al. (2008) | Q | |
| | $8.3\times10^{1}$ | | Modarresi et al. (2007) | Q | 67 |
| | $1.2\times10^{2}$ | | English and Carroll (2001) | Q | 230, 231 |
| | $4.6\times10^{2}$ | | Nirmalakhandan et al. (1997) | Q | |
| | $8.9\times10^{1}$ | | Duchowicz et al. (2020) | ? | 185, 21 |



Table A4.1: Amines (C, H, N) (... continued)

| Substance Formula (Trivial Name) [CAS Registry Number] InChIKey | $H_s^{cp}$ (at $T^\ominus$) $\left[\dfrac{\text{mol}}{\text{m}^3\,\text{Pa}}\right]$ | $\dfrac{\text{d}\ln H_s^{cp}}{\text{d}(1/T)}$ [K] | Reference | Type | Note |
|---|---|---|---|---|---|
| 2-naphthylamine $C_{10}H_9N$ [91-59-8] JBIJLHTVPXGSAM-UHFFFAOYSA-N | $1.2\times10^2$ $1.2\times10^2$ $2.0\times10^1$ $8.0\times10^1$ $1.0\times10^2$ $1.2\times10^2$ $4.5\times10^2$ $1.2\times10^2$ $1.2\times10^2$ | | Abraham et al. (1994a) Keshavarz et al. (2022) Duchowicz et al. (2020) Hilal et al. (2008) Modarresi et al. (2007) English and Carroll (2001) Nirmalakhandan et al. (1997) Duchowicz et al. (2020) HSDB (2015) | R Q Q Q Q Q Q ? ? | 299 67 230, 274 185, 21 419 |
| 1,5-naphthalenediamine $C_{10}H_{10}N_2$ [2243-62-1] KQSABULTKYLFEV-UHFFFAOYSA-N | $1.5\times10^5$ | | HSDB (2015) | Q | 99 |
| phentermine $C_{10}H_{15}N$ [122-09-8] DHHVAGZRUROJKS-UHFFFAOYSA-N | 7.0 | | HSDB (2015) | Q | 99 |
| N,N-diethyl-1,4-benzenediamine $C_{10}H_{16}N_2$ [93-05-0] QNGVNLMMEQUVQK-UHFFFAOYSA-N | $1.9\times10^2$ | | HSDB (2015) | Q | 99 |
| 3,5-diethyltoluene-2,6-diamine $C_{11}H_{18}N_2$ [2095-01-4] RQEOBXYYEPMCPJ-UHFFFAOYSA-N | $6.2\times10^3$ $6.9\times10^3$ $6.1\times10^1$ $2.1\times10^2$ | | Zhang et al. (2010) Zhang et al. (2010) Zhang et al. (2010) Zhang et al. (2010) | Q Q Q Q | 287, 288 287, 289 287, 290 287, 291 |
| 2,4-diethyl-6-methylbenzene-1,3-diamine $C_{11}H_{18}N_2$ [2095-02-5] PISLZQACAJMAIO-UHFFFAOYSA-N | $6.2\times10^3$ $7.0\times10^3$ $6.2\times10^1$ $2.1\times10^2$ | | Zhang et al. (2010) Zhang et al. (2010) Zhang et al. (2010) Zhang et al. (2010) | Q Q Q Q | 287, 288 287, 289 287, 290 287, 291 |
| diphenylamine $C_{12}H_{11}N$ [122-39-4] DMBHHRLKUKUOEG-UHFFFAOYSA-N | 3.7 3.7 $2.9\times10^1$ $2.9\times10^1$ 3.5 1.5 3.5 $4.1\times10^1$ 1.5 3.0 9.4 | | Duchowicz et al. (2020) HSDB (2015) Mackay et al. (2006d) Mackay et al. (1995) Meylan and Howard (1991) Yaws (2003) Howard et al. (1991) Duchowicz et al. (2020) Gharagheizi et al. (2010) Hilal et al. (2008) Meylan and Howard (1991) | V V V V V X X Q Q Q Q | 186 237, 12 412 246 |



Table A4.1: Amines (C, H, N) (... continued)

| Substance<br>Formula<br>(Trivial Name)<br>[CAS Registry Number]<br>InChIKey | $H_s^{cp}$<br>(at $T^\ominus$)<br>$\left[\dfrac{\text{mol}}{\text{m}^3\,\text{Pa}}\right]$ | $\dfrac{\text{d}\ln H_s^{cp}}{\text{d}(1/T)}$<br><br>[K] | Reference | Type | Note |
|---|---|---|---|---|---|
| benzidine<br>$C_{12}H_{12}N_2$<br>[92-87-5]<br>HFACYLZERDEVSX-UHFFFAOYSA-N | $2.2\times10^6$<br>$2.6\times10^5$<br>$2.2\times10^6$<br>$2.5\times10^1$<br>$1.9\times10^5$ | | Mackay et al. (2006d)<br>Lide and Frederikse (1995)<br>Mackay et al. (1995)<br>Mackay et al. (1995)<br>HSDB (2015) | V<br>V<br>V<br>C<br>Q | <br><br><br><br>99 |
| 1,1-diphenylhydrazine<br>$C_{12}H_{12}N_2$<br>[530-50-7]<br>YHYKLKNNBYLTQY-UHFFFAOYSA-N | $2.4\times10^2$ | | HSDB (2015) | Q | 99 |
| 1,2-diphenylhydrazine<br>$C_{12}H_{12}N_2$<br>(N,N'-bianiline)<br>[122-66-7]<br>YBQZXXMEJHZYMB-UHFFFAOYSA-N | $2.1\times10^1$<br>$2.1\times10^1$<br><br>$2.9\times10^3$<br>$2.3\times10^2$ | | Duchowicz et al. (2020)<br>HSDB (2015)<br>Mackay et al. (2006d)<br>Mackay et al. (1995)<br>Duchowicz et al. (2020) | V<br>V<br>V<br>V<br>Q | 186<br><br>558<br><br> |
| 4-(phenylazo)-benzenamine<br>$C_{12}H_{11}N_3$<br>(4-aminoazobenzene)<br>[60-09-3]<br>QPQKUYVSJWQSDY-UHFFFAOYSA-N | $1.1\times10^5$<br>$9.3\times10^2$<br>$1.9\times10^3$<br>$3.2\times10^3$<br>$7.3\times10^5$<br>$3.4\times10^2$ | | HSDB (2015)<br>Gharagheizi et al. (2012)<br>Zhang et al. (2010)<br>Zhang et al. (2010)<br>Zhang et al. (2010)<br>Zhang et al. (2010) | Q<br>Q<br>Q<br>Q<br>Q<br>Q | 99<br><br>287, 288<br>287, 289<br>287, 290<br>287, 291 |
| azobenzene<br>$C_{12}H_{10}N_2$<br>[103-33-3]<br>DMLAVOWQYNRWNQ-UHFFFAOYSA-N | $7.3\times10^{-1}$<br>$7.0\times10^{-1}$<br>$2.6$ | | Duchowicz et al. (2020)<br>HSDB (2015)<br>Duchowicz et al. (2020) | V<br>V<br>Q | 186<br><br> |
| 2-aminobiphenyl<br>$C_{12}H_{11}N$<br>[90-41-5]<br>TWBPWBPGNQWFSJ-UHFFFAOYSA-N | $6.6\times10^1$ | | HSDB (2015) | Q | 99 |
| 4-aminobiphenyl<br>$C_{12}H_{11}N$<br>[92-67-1]<br>DMVOXQPQNTYEKQ-UHFFFAOYSA-N | $6.6\times10^1$ | | HSDB (2015) | Q | 99 |
| N-phenyl-1,4-benzenediamine<br>$C_{12}H_{12}N_2$<br>(p-aminodiphenylamine)<br>[101-54-2]<br>ATGUVEKSASEFFO-UHFFFAOYSA-N | $1.5\times10^3$<br>$2.7\times10^4$<br>$5.9\times10^2$<br>$3.6\times10^3$ | | Yaws (2003)<br>HSDB (2015)<br>Gharagheizi et al. (2012)<br>Gharagheizi et al. (2010) | X<br>Q<br>Q<br>Q | 237<br>99<br><br>246 |
| 2-fluorenamine<br>$C_{13}H_{11}N$<br>[153-78-6]<br>CFRFHWQYWJMEJN-UHFFFAOYSA-N | $2.7\times10^2$ | | HSDB (2015) | Q | 545 |



Table A4.1: Amines (C, H, N) (... continued)

| Substance<br>Formula<br>(Trivial Name)<br>[CAS Registry Number]<br>InChIKey | $H_s^{cp}$<br>(at $T^\ominus$)<br>$\left[\dfrac{\mathrm{mol}}{\mathrm{m^3\,Pa}}\right]$ | $\dfrac{\mathrm{d}\ln H_s^{cp}}{\mathrm{d}(1/T)}$<br><br>[K] | Reference | Type | Note |
|---|---|---|---|---|---|
| 4,4'-methylenebisbenzenamine<br>$C_{13}H_{14}N_2$<br>[101-77-9]<br>YBRVSVVVWCFQMG-UHFFFAOYSA-N | $1.9\times10^5$ | | HSDB (2015) | V | |
| 2-anthracenamine<br>$C_{14}H_{11}N$<br>[613-13-8]<br>YCSBALJAGZKWFF-UHFFFAOYSA-N | $3.3\times10^1$ | | HSDB (2015) | Q | 545 |
| 3,3'-dimethylbenzidine<br>$C_{14}H_{16}N_2$<br>[119-93-7]<br>NUIURNJTPRWVAP-UHFFFAOYSA-N | $1.6\times10^5$ | | HSDB (2015) | Q | 447 |
| N,N-dimethyl-4-(phenylazo)-<br>benzenamine<br>$C_{14}H_{15}N_3$<br>[60-11-7]<br>JCYPECIVGRXBMO-UHFFFAOYSA-N | $1.4\times10^3$<br>$4.2\times10^1$<br>$4.1\times10^1$<br>$8.2\times10^1$<br>$1.0\times10^1$ | | HSDB (2015)<br>Zhang et al. (2010)<br>Zhang et al. (2010)<br>Zhang et al. (2010)<br>Zhang et al. (2010) | V<br>Q<br>Q<br>Q<br>Q | <br>287, 288<br>287, 289<br>287, 290<br>287, 291 |
| *o*-aminoazotoluene<br>$C_{14}H_{15}N_3$<br>[97-56-3]<br>PFRYFZZSECNQOL-UHFFFAOYSA-N | $3.1\times10^2$<br>$3.8\times10^2$ | | Duchowicz et al. (2020)<br>Duchowicz et al. (2020) | V<br>Q | 186 |
| N-ethyl-N-<br>phenylbenzenemethanamine<br>$C_{15}H_{17}N$<br>[92-59-1]<br>HSZCJVZRHXPCIA-UHFFFAOYSA-N | 1.1<br>1.1<br>$4.6\times10^{-1}$<br>6.7 | | Zhang et al. (2010)<br>Zhang et al. (2010)<br>Zhang et al. (2010)<br>Zhang et al. (2010) | Q<br>Q<br>Q<br>Q | 287, 288<br>287, 289<br>287, 290<br>287, 291 |
| N-(1-methylethyl)-N'-phenyl-1,4-<br>benzenediamine<br>$C_{15}H_{18}N_2$<br>(4-(iso-<br>propylamino)diphenylamine)<br>[101-72-4]<br>OUBMGJOQLXMSNT-UHFFFAOYSA-N | $7.0\times10^3$ | | HSDB (2015) | Q | 99 |
| 4,4'-methylene-bis-(N-<br>methylaniline)<br>$C_{15}H_{18}N_2$<br>[1807-55-2]<br>ZMVMYBGDGJLCHV-UHFFFAOYSA-N | $3.4\times10^4$ | | HSDB (2015) | Q | 99 |
| C.I. Food Yellow 10<br>$C_{16}H_{13}N_3$<br>[85-84-7]<br>KLCDQSGLLRINHY-UHFFFAOYSA-N | $1.9\times10^4$ | | HSDB (2015) | Q | 99 |





Table A4.1: Amines (C, H, N) (...continued)

| Substance<br>Formula<br>(Trivial Name)<br>[CAS Registry Number]<br>InChIKey | $H_s^{cp}$<br>(at $T^{\ominus}$)<br>$\left[\dfrac{\text{mol}}{\text{m}^3\,\text{Pa}}\right]$ | $\dfrac{\text{d}\ln H_s^{cp}}{\text{d}(1/T)}$<br><br>[K] | Reference | Type | Note |
|---|---|---|---|---|---|
| 3,3',5,5'-tetramethylbenzidine<br>$C_{16}H_{20}N_2$<br>[54827-17-7]<br>UAIUNKRWKOVEES-UHFFFAOYSA-N | $1.3\times10^5$ | | HSDB (2015) | Q | 99 |
| N-phenyl-1-naphthalenamine<br>$C_{16}H_{13}N$<br>[90-30-2]<br>XQVWYOYUZDUNRW-UHFFFAOYSA-N | $7.0\times10^1$<br>$9.7\times10^1$<br>$4.6\times10^1$<br>$1.2\times10^1$<br>$2.8\times10^2$ | | HSDB (2015)<br>Zhang et al. (2010)<br>Zhang et al. (2010)<br>Zhang et al. (2010)<br>Zhang et al. (2010) | V<br>Q<br>Q<br>Q<br>Q | <br>287, 288<br>287, 289<br>287, 290<br>287, 291 |
| yellow OB<br>$C_{17}H_{15}N_3$<br>[131-79-3]<br>BWLVSYUUKOQICP-FMQUCBEESA-N | $1.8\times10^4$ | | HSDB (2015) | Q | 99 |
| auramine<br>$C_{17}H_{21}N_3$<br>[492-80-8]<br>JPIYZTWMUGTEHX-UHFFFAOYSA-N | $1.2\times10^2$ | | HSDB (2015) | V | |
| benzphetamine<br>$C_{17}H_{21}N$<br>[156-08-1]<br>YXKTVDFXDRQTKV-UHFFFAOYSA-N | $2.3\times10^1$ | | HSDB (2015) | Q | 99 |
| 4,4'-methylenebis(N,N-dimethylbenzenamine)<br>$C_{17}H_{22}N_2$<br>(bis(p-dimethylamino)phenylmethane)<br>[101-61-1]<br>JNRLEMMIVRBKJE-UHFFFAOYSA-N | $8.2\times10^1$ | | HSDB (2015) | Q | 99 |
| phencyclidine<br>$C_{17}H_{25}N$<br>[77-10-1]<br>JTJMJGYZQZDUJJ-UHFFFAOYSA-N | 1.8 | | HSDB (2015) | Q | 99 |
| N,N'-diphenyl-1,4-benzenediamine<br>$C_{18}H_{16}N_2$<br>[74-31-7]<br>UTGQNNCQYDRXCH-UHFFFAOYSA-N | $4.7\times10^4$ | | HSDB (2015) | Q | 99 |
| N-(1,3-dimethylbutyl)-N'-phenyl-1,4-phenylenediamine<br>$C_{18}H_{24}N_2$<br>[793-24-8]<br>ZZMVLMVFYMGSMY-UHFFFAOYSA-N | $2.9\times10^3$<br>$3.9\times10^2$<br>$3.9\times10^1$<br>$2.3\times10^3$ | | Zhang et al. (2010)<br>Zhang et al. (2010)<br>Zhang et al. (2010)<br>Zhang et al. (2010) | Q<br>Q<br>Q<br>Q | 287, 288<br>287, 289<br>287, 290<br>287, 291 |



Table A4.1: Amines (C, H, N) (. . . continued)

| Substance<br>Formula<br>(Trivial Name)<br>[CAS Registry Number]<br>InChIKey | $H_s^{cp}$<br>(at $T^\ominus$)<br>$\left[\dfrac{\mathrm{mol}}{\mathrm{m^3\,Pa}}\right]$ | $\dfrac{\mathrm{d}\ln H_s^{cp}}{\mathrm{d}(1/T)}$<br><br>[K] | Reference | Type | Note |
|---|---|---|---|---|---|
| amitraz<br>$C_{19}H_{23}N_3$<br>[33089-61-1]<br>QXAITBQSYVNQDR-UHFFFAOYSA-N | 1.0<br>1.3<br>$1.4\times10^2$<br>$2.1\times10^1$<br>1.0 | | MacBean (2012b)<br>Keshavarz et al. (2022)<br>Duchowicz et al. (2020)<br>Modarresi et al. (2007)<br>Duchowicz et al. (2020) | X<br>Q<br>Q<br>Q<br>? | 350<br><br><br>67<br>185, 21 |
| N,N'-bis(1-ethyl-3-methylpentyl)-<br>1,4-benzenediamine<br>$C_{22}H_{40}N_2$<br>[139-60-6]<br>JUHXTONDLXIGGK-UHFFFAOYSA-N | $5.8\times10^1$<br><br>5.8<br>1.8<br>$1.9\times10^1$ | | Zhang et al. (2010)<br><br>Zhang et al. (2010)<br>Zhang et al. (2010)<br>Zhang et al. (2010) | Q<br><br>Q<br>Q<br>Q | 287, 288<br><br>287, 289<br>287, 290<br>287, 291 |
| p,p'-benzylidenebis(N,N-<br>dimethylaniline)<br>$C_{23}H_{26}N_2$<br>(leucomalachite green)<br>[129-73-7]<br>WZKXBGJNNCGHIC-UHFFFAOYSA-N | $1.0\times10^3$ | | HSDB (2015) | Q | 99 |
| N-phenyl-N-(2,4,4-trimethyl-2-<br>pentanyl)-1-naphthalenamine<br>$C_{24}H_{29}N$<br>[51772-35-1]<br>SNWVRVDHQRBBFG-UHFFFAOYSA-N | $6.4\times10^{-1}$<br><br>$9.7\times10^{-1}$<br>$9.0\times10^{-1}$<br>$1.1\times10^1$ | | Zhang et al. (2010)<br><br>Zhang et al. (2010)<br>Zhang et al. (2010)<br>Zhang et al. (2010) | Q<br><br>Q<br>Q<br>Q | 287, 288<br><br>287, 289<br>287, 290<br>287, 291 |
| tris(2-ethylhexyl)amine<br>$C_{24}H_{51}N$<br>[1860-26-0]<br>BZUDVELGTZDOIG-UHFFFAOYSA-N | $7.0\times10^{-4}$<br>$1.2\times10^{-2}$<br>$6.1\times10^{-6}$<br>$3.7\times10^{-4}$ | | Zhang et al. (2010)<br>Zhang et al. (2010)<br>Zhang et al. (2010)<br>Zhang et al. (2010) | Q<br>Q<br>Q<br>Q | 287, 288<br>287, 289<br>287, 290<br>287, 291 |
| 4,4',4''-methylidyne-tris(N,N-<br>dimethylbenzenamine)<br>$C_{25}H_{31}N_3$<br>(Leucocrystal violet)<br>[603-48-5]<br>OAZWDJGLIYNYMU-UHFFFAOYSA-N | $6.4\times10^4$<br><br>$3.1\times10^4$<br>$3.5\times10^2$<br>$1.1\times10^6$ | | Zhang et al. (2010)<br><br>Zhang et al. (2010)<br>Zhang et al. (2010)<br>Zhang et al. (2010) | Q<br><br>Q<br>Q<br>Q | 287, 288<br><br>287, 289<br>287, 290<br>287, 291 |
| N-phenylbenzenamide<br>$C_{30}H_{47}N$<br>[68608-79-7]<br>FSPSHPMYFQHGQD-UHFFFAOYSA-N | $8.2\times10^{-2}$<br>$4.7\times10^{-1}$<br>$1.5\times10^{-1}$<br>$2.9\times10^{-1}$ | | Zhang et al. (2010)<br>Zhang et al. (2010)<br>Zhang et al. (2010)<br>Zhang et al. (2010) | Q<br>Q<br>Q<br>Q | 287, 288<br>287, 289<br>287, 290<br>287, 291 |
| 4,4',4''-methanetriyltris(N,N-<br>diethylaniline)<br>$C_{31}H_{43}N_3$<br>[68814-02-8]<br>HOGMPEULJBVHLZ-UHFFFAOYSA-N | $9.0\times10^4$<br><br>$7.0\times10^5$<br>$1.7\times10^3$<br>$1.5\times10^6$ | | Zhang et al. (2010)<br><br>Zhang et al. (2010)<br>Zhang et al. (2010)<br>Zhang et al. (2010) | Q<br><br>Q<br>Q<br>Q | 287, 288<br><br>287, 289<br>287, 290<br>287, 291 |



## A4.2   Heterocycles with nitrogen (C, H, N)

Table A4.2: Heterocycles with nitrogen (C, H, N)

| Substance Formula (Trivial Name) [CAS Registry Number] InChIKey | $H_s^{cp}$ (at $T^\ominus$) $\left[\dfrac{\mathrm{mol}}{\mathrm{m^3\,Pa}}\right]$ | $\dfrac{\mathrm{d}\ln H_s^{cp}}{\mathrm{d}(1/T)}$ [K] | Reference | Type | Note |
|---|---|---|---|---|---|
| amitrole $C_2H_4N_4$ [61-82-5] KLSJWNVTNUYHDU-UHFFFAOYSA-N | $4.5\times10^7$ $4.5\times10^7$ $6.1\times10^9$ $1.5\times10^3$ | | Duchowicz et al. (2020) HSDB (2015) Mackay et al. (2006d) Duchowicz et al. (2020) | V V V Q | 186 |
| azetidine $C_3H_7N$ [503-29-7] HONIICLYMWZJFZ-UHFFFAOYSA-N | 4.8 | | Ebert et al. (2023) | ? | 365 |
| imidazole $C_3H_4N_2$ [288-32-4] RAXXELZNTBOGNW-UHFFFAOYSA-N | $3.3\times10^3$ $2.1\times10^3$ $3.1\times10^3$ | | Du et al. (2017) Du et al. (2017) Du et al. (2017) | M Q Q | 478 549 |
| 1,3,5-triazine-2,4,6-triamine $C_3H_6N_6$ [108-78-1] JDSHMPZPIAZGSV-UHFFFAOYSA-N | $5.5\times10^8$ $5.2\times10^7$ $6.7\times10^8$ $5.8\times10^9$ $8.4\times10^8$ | | HSDB (2015) Zhang et al. (2010) Zhang et al. (2010) Zhang et al. (2010) Zhang et al. (2010) | V Q Q Q Q | 287, 288 287, 289 287, 290 287, 291 |
| pyrrolidine $C_4H_8NH$ [123-75-1] RWRDLPDLKQPQOW-UHFFFAOYSA-N | 4.2 4.2 1.8 $1.3\times10^1$ 6.0 $5.3\times10^{-1}$ 4.1 $8.4\times10^{-1}$ 2.0 4.1 | 7600 | Amoore and Buttery (1978) Cabani et al. (1971a) Keshavarz et al. (2022) Duchowicz et al. (2020) Hilal et al. (2008) Modarresi et al. (2007) Yaffe et al. (2003) Katritzky et al. (1998) Suzuki et al. (1992) Duchowicz et al. (2020) | V T Q Q Q Q Q Q Q ? | 299 67 248, 249 232 185, 21 |
| 1-pyrroline $C_4H_7N$ [5724-81-2] ZVJHJDDKYZXRJI-UHFFFAOYSA-N | 1.6 | | Amoore and Buttery (1978) | M | |
| 3-pyrroline $C_4H_7N$ [109-96-6] JVQIKJMSUIMUDI-UHFFFAOYSA-N | 4.9 | | Amoore and Buttery (1978) | V | |
| 1,4-diazacyclohexane $C_4H_{10}N_2$ (piperazine) [110-85-0] GLUUGHFHXGJENI-UHFFFAOYSA-N | $4.5\times10^3$ $1.0\times10^2$ | 7400 11000 | Nguyen (2013) Cabani et al. (1975a) | M T | 11 |





Table A4.2: Heterocycles with nitrogen (C, H, N) (...continued)

| Substance<br>Formula<br>(Trivial Name)<br>[CAS Registry Number]<br>InChIKey | $H_s^{cp}$<br>(at $T^{\ominus}$)<br>$\left[\dfrac{\mathrm{mol}}{\mathrm{m^3\,Pa}}\right]$ | $\dfrac{\mathrm{d}\ln H_s^{cp}}{\mathrm{d}(1/T)}$<br>[K] | Reference | Type | Note |
|---|---|---|---|---|---|
| pyrrole<br>$C_4H_5N$<br>(1H-pyrrole)<br>[109-97-7]<br>KAESVJOAVNADME-UHFFFAOYSA-N | $5.5\times10^{-1}$<br>$6.1\times10^{-1}$<br>$6.1\times10^{-1}$<br>$1.8$<br>$4.1$<br>$7.2\times10^{-1}$<br>$8.6\times10^{-1}$<br>$5.4\times10^{-1}$<br>$4.2$<br>$5.5\times10^{-1}$ | | Hawthorne et al. (1985)<br>Mackay et al. (2006d)<br>Mackay et al. (1995)<br>Keshavarz et al. (2022)<br>Duchowicz et al. (2020)<br>Hilal et al. (2008)<br>Modarresi et al. (2007)<br>Yaffe et al. (2003)<br>Katritzky et al. (1998)<br>Duchowicz et al. (2020) | M<br>V<br>V<br>Q<br>Q<br>Q<br>Q<br>Q<br>Q<br>? | <br><br><br><br>184<br><br>67<br>248, 249<br><br>185, 21 |
| 1-methyl-1H-imidazole<br>$C_4H_6N_2$<br>[616-47-7]<br>MCTWTZJPVLRJOU-UHFFFAOYSA-N | $6.9\times10^{1}$<br>$3.4\times10^{2}$<br>$8.9\times10^{1}$<br>$1.1\times10^{2}$ | | Du et al. (2017)<br>Du et al. (2017)<br>Du et al. (2017)<br>Hilal et al. (2008) | M<br>Q<br>Q<br>Q | 478<br>549<br><br> |
| 1,3-diazine<br>$C_4H_4N_2$<br>[289-95-2]<br>CZPWVGJYEJSRLH-UHFFFAOYSA-N | $1.0\times10^{1}$ | | Hilal et al. (2008) | Q | |
| N-methylpyrrolidine<br>$C_4H_8NCH_3$<br>[120-94-5]<br>AVFZOVWCLRSYKC-UHFFFAOYSA-N | $3.3\times10^{-1}$<br>$1.7\times10^{-1}$<br>$1.2$<br>$2.2\times10^{-1}$<br>$5.6\times10^{-2}$<br>$3.2\times10^{-1}$<br>$3.3\times10^{-1}$ | 7600 | Cabani et al. (1971a)<br>Keshavarz et al. (2022)<br>Duchowicz et al. (2020)<br>Hilal et al. (2008)<br>Modarresi et al. (2007)<br>Suzuki et al. (1992)<br>Duchowicz et al. (2020) | T<br>Q<br>Q<br>Q<br>Q<br>Q<br>? | <br><br>184<br><br>67<br>232<br>185, 21 |
| piperidine<br>$C_5H_{10}NH$<br>[110-89-4]<br>NQRYJNQNLNOLGT-UHFFFAOYSA-N | $2.8$<br>$2.0$<br>$2.2$<br>$2.5$<br>$1.3\times10^{1}$<br>$7.3$<br>$1.0$<br>$7.2\times10^{-1}$<br>$6.7\times10^{-1}$<br>$1.5$<br>$2.2$ | 7900<br><br>7900 | Bernauer and Dohnal (2009)<br>Amoore and Buttery (1978)<br>Cabani et al. (1971a)<br>Keshavarz et al. (2022)<br>Duchowicz et al. (2020)<br>Hilal et al. (2008)<br>Modarresi et al. (2007)<br>Katritzky et al. (1998)<br>Russell et al. (1992)<br>Suzuki et al. (1992)<br>Duchowicz et al. (2020) | M<br>V<br>T<br>Q<br>Q<br>Q<br>Q<br>Q<br>Q<br>Q<br>? | <br><br><br><br>299<br><br>67<br><br>279<br>232<br>185, 21 |
| 2-ethylimidazole<br>$C_5H_8N_2$<br>[1072-62-4]<br>PQAMFDRRWURCFQ-UHFFFAOYSA-N | $7.6\times10^{2}$<br>$1.4\times10^{4}$<br>$8.2\times10^{2}$ | | Du et al. (2017)<br>Du et al. (2017)<br>Du et al. (2017) | M<br>Q<br>Q | 478<br>549<br> |
| 1,2-dimethylimidazole<br>$C_5H_8N_2$<br>[1739-84-0]<br>GIWQSPITLQVMSG-UHFFFAOYSA-N | $4.3\times10^{1}$<br>$2.9\times10^{2}$<br>$3.3\times10^{1}$ | | Du et al. (2017)<br>Du et al. (2017)<br>Du et al. (2017) | M<br>Q<br>Q | 478<br>549<br> |



Table A4.2: Heterocycles with nitrogen (C, H, N) (...continued)

| Substance<br>Formula<br>(Trivial Name)<br>[CAS Registry Number]<br>InChIKey | $H_s^{cp}$<br>(at $T^\ominus$)<br>$\left[\dfrac{\text{mol}}{\text{m}^3\,\text{Pa}}\right]$ | $\dfrac{\text{d}\ln H_s^{cp}}{\text{d}(1/T)}$<br><br>[K] | Reference | Type | Note |
|---|---|---|---|---|---|
| N-methylpiperazine<br>$C_5H_{12}N_2$<br>(1-methylpiperazine)<br>[109-01-3]<br>PVOAHINGSUIXLS-UHFFFAOYSA-N | $1.8\times10^3$<br>$2.0\times10^2$ | 9900<br>11000 | Nguyen (2013)<br>Cabani et al. (1975a) | M<br>T | 11 |
| 1-methyl-1H-pyrrole<br>$C_5H_7N$<br>[96-54-8]<br>OXHNLMTVIGZXSG-UHFFFAOYSA-N | $9.0\times10^{-3}$ | | Hilal et al. (2008) | Q | |
| pyridine<br>$C_5H_5N$<br>[110-86-1]<br>JUJWROOIHBZHMG-UHFFFAOYSA-N | $9.1\times10^{-1}$<br>1.1<br>$4.6\times10^{-2}$<br>$9.8\times10^{-1}$<br>$5.5\times10^{-1}$<br>$8.2\times10^{-1}$<br>1.1<br>$7.1\times10^{-1}$<br>1.1<br>$8.4\times10^{-1}$<br>$2.4\times10^{-1}$<br>1.1<br>$7.5\times10^{-1}$<br>$6.4\times10^{-1}$<br><br>1.2<br>1.3<br>1.2<br>$9.0\times10^{-2}$<br>1.8<br>3.3<br>1.2<br>$9.0\times10^{-1}$<br>1.1<br><br>1.1<br>$8.9\times10^{-1}$<br>1.1 | 4700<br>6000<br>-2300<br><br><br><br><br><br>5900<br><br><br><br><br><br>6000<br><br><br><br><br><br><br><br><br><br><br>5400 | Brockbank (2013)<br>Bernauer and Dohnal (2009)<br>Dewulf et al. (1999)<br>Welke et al. (1998)<br>Chaintreau et al. (1995)<br>Hawthorne et al. (1985)<br>Arnett and Chawla (1979)<br>Amoore and Buttery (1978)<br>Andon et al. (1954)<br>Keshavarz et al. (2022)<br>Duchowicz et al. (2020)<br>Li et al. (2014)<br>Hilal et al. (2008)<br>Modarresi et al. (2007)<br>Kühne et al. (2005)<br>Yaffe et al. (2003)<br>Yao et al. (2002)<br>English and Carroll (2001)<br>Katritzky et al. (1998)<br>Nirmalakhandan et al. (1997)<br>Russell et al. (1992)<br>Suzuki et al. (1992)<br>Duchowicz et al. (2020)<br>Mackay et al. (2006d)<br>Kühne et al. (2005)<br>Yaws (1999)<br>Yaws and Yang (1992)<br>Abraham et al. (1990)<br>Staudinger and Roberts (2001) | L<br>M<br>M<br>M<br>M<br>M<br>M<br>M<br>M<br>Q<br>Q<br>Q<br>Q<br>Q<br>Q<br>Q<br>Q<br>Q<br>Q<br>Q<br>Q<br>Q<br>?<br>?<br>?<br>?<br>?<br>?<br>W | 1<br><br><br><br><br><br>559<br><br>336<br><br>299<br>241<br><br>67<br><br>248, 249<br>229<br>230, 231<br><br><br>279<br>232<br>185, 21<br><br><br>21<br>21<br><br>560 |
| pyridine-d5<br>$C_5D_5N$<br>[7291-22-7]<br>JUJWROOIHBZHMG-RALIUCGRSA-N | 4.2 | 10000 | Hiatt (2013) | M | |



Table A4.2: Heterocycles with nitrogen (C, H, N) (. . . continued)

| Substance Formula (Trivial Name) [CAS Registry Number] InChIKey | $H_s^{cp}$ (at $T^{\ominus}$) $\left[\dfrac{\text{mol}}{\text{m}^3\,\text{Pa}}\right]$ | $\dfrac{\text{d}\ln H_s^{cp}}{\text{d}(1/T)}$ [K] | Reference | Type | Note |
|---|---|---|---|---|---|
| 4-aminopyridine $C_5H_6N_2$ [504-24-5] NUKYPUAOHBNCPY-UHFFFAOYSA-N | $4.3\times10^4$ | | HSDB (2015) | V | |
| 2-aminopyridine $C_5H_6N_2$ [504-29-0] ICSNLGPSRYBMBD-UHFFFAOYSA-N | $3.9\times10^3$ | | HSDB (2015) | Q | 99 |
| 2-methylpyrazine $C_4N_2H_3CH_3$ [109-08-0] CAWHJQAVHZEVTJ-UHFFFAOYSA-N | 4.5 4.9 $6.8\times10^{-1}$ 4.8 4.9 $1.6\times10^2$ 3.1 1.2 4.5 | | Buttery et al. (1971) Keshavarz et al. (2022) Duchowicz et al. (2020) Hilal et al. (2008) Modarresi et al. (2007) Katritzky et al. (1998) Nirmalakhandan et al. (1997) Russell et al. (1992) Duchowicz et al. (2020) | M Q Q Q Q Q Q Q ? | 184 67 279 185, 21 |
| adenine $C_5H_5N_5$ [73-24-5] GFFGJBXGBJISGV-UHFFFAOYSA-N | $1.3\times10^9$ $5.3\times10^5$ | | Duchowicz et al. (2020) Duchowicz et al. (2020) | V Q | 186 |
| 9-methyladenine $C_6H_7N_5$ [700-00-5] WRXCXOUDSPTXNX-UHFFFAOYSA-N | $3.9\times10^6$ | | Ebert et al. (2023) | ? | 365 |
| N-methylpiperidine $C_5H_{10}NCH_3$ [626-67-5] PAMIQIKDUOTOBW-UHFFFAOYSA-N | $2.4\times10^{-1}$ $2.9\times10^{-1}$ $2.3\times10^{-1}$ 1.2 $4.8\times10^{-1}$ $1.3\times10^{-1}$ $3.0\times10^{-1}$ $2.9\times10^{-1}$ $2.2\times10^{-1}$ $9.9\times10^{-2}$ $2.5\times10^{-1}$ $2.9\times10^{-1}$ | 7900 6300 6600 | Abraham et al. (1994a) Cabani et al. (1971a) Keshavarz et al. (2022) Duchowicz et al. (2020) Hilal et al. (2008) Modarresi et al. (2007) Kühne et al. (2005) English and Carroll (2001) Katritzky et al. (1998) Nirmalakhandan et al. (1997) Russell et al. (1992) Suzuki et al. (1992) Duchowicz et al. (2020) Kühne et al. (2005) | R T Q Q Q Q Q Q Q Q Q Q ? ? | 67 230, 231 279 232 185, 21 |
| triethylenediamine $C_6H_{12}N_2$ [280-57-9] IMNIMPAHZVJRPE-UHFFFAOYSA-N | $3.1\times10^3$ $8.9\times10^2$ $2.2\times10^3$ | | Du et al. (2017) Du et al. (2017) Du et al. (2017) | M Q Q | 478 549 |





Table A4.2: Heterocycles with nitrogen (C, H, N) (...continued)

| Substance Formula (Trivial Name) [CAS Registry Number] InChIKey | $H_s^{cp}$ (at $T^\ominus$) $\left[\dfrac{\mathrm{mol}}{\mathrm{m}^3\,\mathrm{Pa}}\right]$ | $\dfrac{\mathrm{d}\ln H_s^{cp}}{\mathrm{d}(1/T)}$ [K] | Reference | Type | Note |
|---|---|---|---|---|---|
| 2-ethyl-4-methylimidazole $C_6H_{10}N_2$ [931-36-2] ULKLGIFJWFIQFF-UHFFFAOYSA-N | $2.8\times10^2$ $1.4\times10^3$ $3.1\times10^2$ | | Du et al. (2017) Du et al. (2017) Du et al. (2017) | M Q Q | 478 549 |
| 3-(aminomethyl)pyridine $C_6H_8N_2$ [3731-52-0] HDOUGSFASVGDCS-UHFFFAOYSA-N | $4.3\times10^2$ $5.0\times10^2$ $3.7\times10^2$ | | Du et al. (2017) Du et al. (2017) Du et al. (2017) | M Q Q | 478 549 |
| N,N'-dimethylpiperazine $C_6H_{14}N_2$ (1,4-dimethylpiperazine) [106-58-1] RXYPXQSKLGGKOL-UHFFFAOYSA-N | $2.0\times10^2$ $1.4\times10^2$ | 11000 11000 | Nguyen (2013) Cabani et al. (1975a) | M T | 11 |
| cyromazine $C_6H_{10}N_6$ [66215-27-8] LVQDKIWDGQRHTE-UHFFFAOYSA-N | $1.7\times10^8$ $1.7\times10^8$ $1.3\times10^8$ | | Duchowicz et al. (2020) HSDB (2015) Duchowicz et al. (2020) | V V Q | 186 |
| 1H-benzotriazole $C_6H_5N_3$ (1,2,3-benzotriazole) [95-14-7] QRUDEWIWKLJBPS-UHFFFAOYSA-N | $3.1\times10^1$ | | HSDB (2015) | V | |
| 1-piperazineethanamine $C_6H_{15}N_3$ (N-(2-aminoethyl)piperazine) [140-31-8] IMUDHTPIFIBORV-UHFFFAOYSA-N | $1.5\times10^7$ | | HSDB (2015) | Q | 99 |
| 2-methylpyridine $C_5H_4NCH_3$ (2-picoline; $\alpha$-picoline) [109-06-8] BSKHPKMHTQYZBB-UHFFFAOYSA-N | $9.8\times10^{-1}$ $9.9\times10^{-1}$ $1.1$ $8.0\times10^{-2}$ $4.1\times10^{-1}$ $1.2$ $1.0$ $1.0$ $1.4\times10^2$ $2.1$ $1.3$ $8.8\times10^{-1}$ $9.9\times10^{-1}$ $9.9\times10^{-1}$ $9.9\times10^{-1}$ | 6300 6400 6400 6300 | Brockbank (2013) Andon et al. (1954) Keshavarz et al. (2022) Duchowicz et al. (2020) Hilal et al. (2008) Modarresi et al. (2007) Kühne et al. (2005) Yaffe et al. (2003) Yao et al. (2002) English and Carroll (2001) Katritzky et al. (1998) Nirmalakhandan et al. (1997) Suzuki et al. (1992) Duchowicz et al. (2020) Mackay et al. (2006d) Kühne et al. (2005) Yaws (1999) | L M Q Q Q Q Q Q Q Q Q Q Q ? ? ? ? | 1 336 67 248, 249 229 230, 231 232 185, 21 21 |



Table A4.2: Heterocycles with nitrogen (C, H, N) (. . . continued)

| Substance Formula (Trivial Name) [CAS Registry Number] InChIKey | $H_s^{cp}$ (at $T^\ominus$) $\left[\dfrac{\text{mol}}{\text{m}^3\,\text{Pa}}\right]$ | $\dfrac{\text{d}\ln H_s^{cp}}{\text{d}(1/T)}$ [K] | Reference | Type | Note |
|---|---|---|---|---|---|
| | $3.4\times10^{-1}$ | | Yaws and Yang (1992) | ? | 21 |
| | $9.9\times10^{-1}$ | | Abraham et al. (1990) | ? | |
| | | | Staudinger and Roberts (2001) | W | 560 |
| 3-methylpyridine | 1.2 | 6400 | Brockbank (2013) | L | 1 |
| $C_5H_4NCH_3$ | $4.2\times10^{-1}$ | | Chaintreau et al. (1995) | M | |
| (3-picoline; $\beta$-picoline) | 1.3 | 6300 | Andon et al. (1954) | M | 336 |
| [108-99-6] | 1.1 | | Keshavarz et al. (2022) | Q | |
| ITQTTZVARXURQS-UHFFFAOYSA-N | $8.0\times10^{-2}$ | | Duchowicz et al. (2020) | Q | 299 |
| | $8.8\times10^{-1}$ | | Hilal et al. (2008) | Q | |
| | $8.5\times10^{-1}$ | | Modarresi et al. (2007) | Q | 67 |
| | | 6400 | Kühne et al. (2005) | Q | |
| | 1.0 | | Yaffe et al. (2003) | Q | 248, 272 |
| | 1.3 | | Yao et al. (2002) | Q | 229, 267 |
| | $9.5\times10^{-1}$ | | English and Carroll (2001) | Q | 230, 231 |
| | 2.3 | | Katritzky et al. (1998) | Q | |
| | 1.3 | | Nirmalakhandan et al. (1997) | Q | |
| | $8.8\times10^{-1}$ | | Suzuki et al. (1992) | Q | 232 |
| | 1.3 | | Duchowicz et al. (2020) | ? | 185, 21 |
| | 1.3 | | Mackay et al. (2006d) | ? | |
| | | 6300 | Kühne et al. (2005) | ? | |
| | 1.4 | | Yaws (1999) | ? | 21 |
| | $5.4\times10^{-1}$ | | Yaws and Yang (1992) | ? | 21 |
| | 1.3 | | Abraham et al. (1990) | ? | |
| | | | Staudinger and Roberts (2001) | W | 560 |
| 4-methylpyridine | 1.6 | 6500 | Brockbank (2013) | L | 1 |
| $C_5H_4NCH_3$ | 1.7 | 6500 | Andon et al. (1954) | M | 336 |
| [108-89-4] | 1.1 | | Keshavarz et al. (2022) | Q | |
| FKNQCJSGGFJEIZ-UHFFFAOYSA-N | $8.0\times10^{-2}$ | | Duchowicz et al. (2020) | Q | 184 |
| | 1.6 | | Li et al. (2014) | Q | 241 |
| | $9.0\times10^{-1}$ | | Hilal et al. (2008) | Q | |
| | $8.6\times10^{-1}$ | | Modarresi et al. (2007) | Q | 67 |
| | | 6400 | Kühne et al. (2005) | Q | |
| | 1.2 | | Yaffe et al. (2003) | Q | 248, 272 |
| | 1.2 | | Yao et al. (2002) | Q | 229 |
| | 1.1 | | English and Carroll (2001) | Q | 230, 231 |
| | 2.4 | | Katritzky et al. (1998) | Q | |
| | 1.3 | | Nirmalakhandan et al. (1997) | Q | |
| | 2.1 | | Russell et al. (1992) | Q | 279 |
| | $8.8\times10^{-1}$ | | Suzuki et al. (1992) | Q | 232 |
| | 1.6 | | Duchowicz et al. (2020) | ? | 185, 21 |
| | 1.7 | | Mackay et al. (2006d) | ? | |
| | | 6500 | Kühne et al. (2005) | ? | |
| | 1.7 | | Yaws (1999) | ? | 21, 12 |
| | 1.6 | | Abraham et al. (1990) | ? | |
| | 1.4 | | Arnett and Chawla (1979) | ? | 559 |
| | | | Staudinger and Roberts (2001) | W | 560 |





Table A4.2: Heterocycles with nitrogen (C, H, N) (...continued)

| Substance<br>Formula<br>(Trivial Name)<br>[CAS Registry Number]<br>InChIKey | $H_s^{cp}$<br>(at $T^\ominus$)<br>$\left[\dfrac{\mathrm{mol}}{\mathrm{m}^3\,\mathrm{Pa}}\right]$ | $\dfrac{\mathrm{d}\ln H_s^{cp}}{\mathrm{d}(1/T)}$<br><br>[K] | Reference | Type | Note |
|---|---|---|---|---|---|
| 3-cyanopyridine<br>$C_6H_4N_2$<br>[100-54-9]<br>GZPHSAQLYPIAIN-UHFFFAOYSA-N | $3.6\times10^1$<br>$1.9\times10^1$<br>$5.4$<br>$3.6\times10^1$<br>$1.6\times10^1$<br>$1.1\times10^1$<br>$6.9\times10^1$<br>$1.2\times10^2$<br>$3.6\times10^1$ | | Abraham et al. (1994a)<br>Keshavarz et al. (2022)<br>Duchowicz et al. (2020)<br>HSDB (2015)<br>Hilal et al. (2008)<br>Modarresi et al. (2007)<br>English and Carroll (2001)<br>Nirmalakhandan et al. (1997)<br>Duchowicz et al. (2020) | R<br>Q<br>Q<br>Q<br>Q<br>Q<br>Q<br>Q<br>? | <br><br><br>99<br><br>67<br>230, 231<br><br>185, 21 |
| 4-cyanopyridine<br>$C_6H_4N_2$<br>[100-48-1]<br>GPHQHTOMRSGBNZ-UHFFFAOYSA-N | $1.1\times10^1$<br>$1.9\times10^1$<br>$5.4$<br>$1.7\times10^1$<br>$1.0\times10^1$<br>$1.2\times10^2$<br>$1.1\times10^1$ | | Abraham et al. (1994a)<br>Keshavarz et al. (2022)<br>Duchowicz et al. (2020)<br>Hilal et al. (2008)<br>Modarresi et al. (2007)<br>Nirmalakhandan et al. (1997)<br>Duchowicz et al. (2020) | R<br>Q<br>Q<br>Q<br>Q<br>Q<br>? | <br><br><br><br>67<br><br>185, 21 |
| 2-ethylpyrazine<br>$C_4N_2H_3(C_2H_5)$<br>[13925-00-3]<br>KVFIJIWMDBAGDP-UHFFFAOYSA-N | $4.0$<br>$6.5$<br>$7.3\times10^{-1}$<br>$2.7$<br>$3.4$<br>$2.7$<br>$4.0$ | | Buttery et al. (1971)<br>Keshavarz et al. (2022)<br>Duchowicz et al. (2020)<br>Hilal et al. (2008)<br>Modarresi et al. (2007)<br>Nirmalakhandan et al. (1997)<br>Duchowicz et al. (2020) | M<br>Q<br>Q<br>Q<br>Q<br>Q<br>? | <br><br>184<br><br>67<br><br>185, 21 |
| 2,5-dimethylpyrazine<br>$C_6H_8N_2$<br>[123-32-0]<br>LCZUOKDVTBMCMX-UHFFFAOYSA-N | $7.1$<br>$5.5$<br>$5.5$<br>$6.4$ | | Marin et al. (1999)<br>Druaux et al. (1998)<br>Marin et al. (1999)<br>Marin et al. (1999) | M<br>M<br>V<br>Q | |
| 2,6-dimethylpyrazine<br>$C_6H_8N_2$<br>(3,5-dimethylpyrazine)<br>[108-50-9]<br>HJFZAYHYIWGLNL-UHFFFAOYSA-N | $9.8\times10^{-1}$ | | Chaintreau et al. (1995) | M | |
| N-ethylpiperidine<br>$C_7H_{15}N$<br>(1-ethylpiperidine)<br>[766-09-6]<br>HTLZVHNRZJPSMI-UHFFFAOYSA-N | $3.9\times10^{-1}$ | <br>6600<br>6600 | Hilal et al. (2008)<br>Kühne et al. (2005)<br>Kühne et al. (2005) | Q<br>Q<br>? | |
| 1H-benzimidazole<br>$C_7H_6N_2$<br>[51-17-2]<br>HYZJCKYKOHLVJF-UHFFFAOYSA-N | $2.7\times10^1$ | | HSDB (2015) | Q | 99 |



Table A4.2: Heterocycles with nitrogen (C, H, N) (...continued)

| Substance Formula (Trivial Name) [CAS Registry Number] InChIKey | $H_s^{cp}$ (at $T^{\ominus}$) $\left[\dfrac{\mathrm{mol}}{\mathrm{m^3\,Pa}}\right]$ | $\dfrac{\mathrm{d}\ln H_s^{cp}}{\mathrm{d}(1/T)}$ [K] | Reference | Type | Note |
|---|---|---|---|---|---|
| 2-ethenylpyridine | $7.0\times10^{-1}$ | | Duchowicz et al. (2020) | V | 186 |
| $C_7H_7N$ | $2.8\times10^{-1}$ | | Duchowicz et al. (2020) | Q | |
| (2-vinylpyridine) | 2.7 | | HSDB (2015) | Q | 99 |
| [100-69-6] | | | | | |
| KGIGUEBEKRSTEW-UHFFFAOYSA-N | | | | | |
| 4-ethenylpyridine | 1.2 | | Duchowicz et al. (2020) | V | 186 |
| $C_7H_7N$ | $2.8\times10^{-1}$ | | Duchowicz et al. (2020) | Q | |
| (4-vinylpyridine) | 3.1 | | HSDB (2015) | Q | 99 |
| [100-43-6] | | | | | |
| KFDVPJUYSDEJTH-UHFFFAOYSA-N | | | | | |
| 2-ethylpyridine | $6.0\times10^{-1}$ | 6700 | Andon et al. (1954) | M | 336 |
| $C_5H_4NC_2H_5$ | 1.5 | | Keshavarz et al. (2022) | Q | |
| [100-71-0] | $8.2\times10^{-2}$ | | Duchowicz et al. (2020) | Q | 299 |
| NRGGMCIBEHEAIL-UHFFFAOYSA-N | $2.9\times10^{-1}$ | | Hilal et al. (2008) | Q | |
| | $6.5\times10^{-1}$ | | Modarresi et al. (2007) | Q | 67 |
| | | 6700 | Kühne et al. (2005) | Q | |
| | $6.1\times10^{-1}$ | | Yaffe et al. (2003) | Q | 248, 249 |
| | 2.4 | | Katritzky et al. (1998) | Q | |
| | 1.1 | | Nirmalakhandan et al. (1997) | Q | |
| | 1.4 | | Suzuki et al. (1992) | Q | 232 |
| | $6.0\times10^{-1}$ | | Duchowicz et al. (2020) | ? | 185, 21 |
| | | 7900 | Kühne et al. (2005) | ? | |
| | $6.0\times10^{-1}$ | | Abraham et al. (1990) | ? | |
| | | | Staudinger and Roberts (2001) | W | 560 |
| 3-ethylpyridine | $9.5\times10^{-1}$ | 6400 | Andon et al. (1954) | M | 336 |
| $C_5H_4NC_2H_5$ | 1.5 | | Keshavarz et al. (2022) | Q | |
| [536-78-7] | $8.2\times10^{-2}$ | | Duchowicz et al. (2020) | Q | |
| MFEIKQPHQINPRI-UHFFFAOYSA-N | $6.7\times10^{-1}$ | | Hilal et al. (2008) | Q | |
| | $6.0\times10^{-1}$ | | Modarresi et al. (2007) | Q | 67 |
| | | 6700 | Kühne et al. (2005) | Q | |
| | 1.0 | | English and Carroll (2001) | Q | 230, 231 |
| | 2.9 | | Katritzky et al. (1998) | Q | |
| | 1.1 | | Nirmalakhandan et al. (1997) | Q | |
| | 1.4 | | Suzuki et al. (1992) | Q | 232 |
| | $9.5\times10^{-1}$ | | Duchowicz et al. (2020) | ? | 185, 21 |
| | | 6200 | Kühne et al. (2005) | ? | |
| | $9.5\times10^{-1}$ | | Abraham et al. (1990) | ? | |
| | | | Staudinger and Roberts (2001) | W | 560 |
| 4-ethylpyridine | 1.2 | 6300 | Andon et al. (1954) | M | 336 |
| $C_5H_4NC_2H_5$ | 1.5 | | Keshavarz et al. (2022) | Q | |
| [536-75-4] | $8.2\times10^{-2}$ | | Duchowicz et al. (2020) | Q | 299 |
| VJXRKZJMGVSXPX-UHFFFAOYSA-N | 1.2 | | Li et al. (2014) | Q | 241 |
| | $7.0\times10^{-1}$ | | Hilal et al. (2008) | Q | |
| | $5.7\times10^{-1}$ | | Modarresi et al. (2007) | Q | 67 |
| | | 6700 | Kühne et al. (2005) | Q | |



Table A4.2: Heterocycles with nitrogen (C, H, N) (. . . continued)

| Substance Formula (Trivial Name) [CAS Registry Number] InChIKey | $H_s^{cp}$ (at $T^{\ominus}$) $\left[\dfrac{\text{mol}}{\text{m}^3\,\text{Pa}}\right]$ | $\dfrac{\text{d}\ln H_s^{cp}}{\text{d}(1/T)}$ [K] | Reference | Type | Note |
|---|---|---|---|---|---|
| | 1.2 | | Yaffe et al. (2003) | Q | 248, 249 |
| | 1.0 | | English and Carroll (2001) | Q | 230, 260 |
| | 2.9 | | Katritzky et al. (1998) | Q | |
| | 1.1 | | Nirmalakhandan et al. (1997) | Q | |
| | 1.4 | | Suzuki et al. (1992) | Q | 232 |
| | 1.2 | | Duchowicz et al. (2020) | ? | 185, 21 |
| | | 6300 | Kühne et al. (2005) | ? | |
| | 1.2 | | Abraham et al. (1990) | ? | |
| | | | Staudinger and Roberts (2001) | W | 560 |
| 2,3-dimethylpyridine $C_5H_3N(CH_3)_2$ (2,3-lutidine) [583-61-9] HPYNZHMRTTWQTB-UHFFFAOYSA-N | 1.4 | 6900 | Andon et al. (1954) | M | 336 |
| | 1.5 | | Keshavarz et al. (2022) | Q | |
| | $2.7\times10^{-2}$ | | Duchowicz et al. (2020) | Q | 184 |
| | $6.2\times10^{-1}$ | | Hilal et al. (2008) | Q | |
| | $2.1\times10^{-1}$ | | Modarresi et al. (2007) | Q | 67 |
| | | 6200 | Kühne et al. (2005) | Q | |
| | 1.2 | | Yaffe et al. (2003) | Q | 248, 272 |
| | 1.6 | | English and Carroll (2001) | Q | 230, 274 |
| | 3.0 | | Katritzky et al. (1998) | Q | |
| | $9.5\times10^{-1}$ | | Nirmalakhandan et al. (1997) | Q | |
| | 1.3 | | Suzuki et al. (1992) | Q | 232 |
| | 1.4 | | Duchowicz et al. (2020) | ? | 185, 21 |
| | 1.4 | | Mackay et al. (2006d) | ? | |
| | | 5800 | Kühne et al. (2005) | ? | |
| | 1.4 | | Abraham et al. (1990) | ? | |
| | | | Staudinger and Roberts (2001) | W | 560 |
| 2,4-dimethylpyridine $C_5H_3N(CH_3)_2$ (2,4-lutidine) [108-47-4] JYYNAJVZFGKDEQ-UHFFFAOYSA-N | $9.9\times10^{-1}$ | | Hawthorne et al. (1985) | M | |
| | 1.5 | 7100 | Andon et al. (1954) | M | 336 |
| | 1.5 | | Keshavarz et al. (2022) | Q | |
| | $2.7\times10^{-2}$ | | Duchowicz et al. (2020) | Q | |
| | 1.5 | | Li et al. (2014) | Q | 241 |
| | $5.1\times10^{-1}$ | | Hilal et al. (2008) | Q | |
| | $2.9\times10^{-1}$ | | Modarresi et al. (2007) | Q | 67 |
| | | 6700 | Kühne et al. (2005) | Q | |
| | 1.5 | | Yaffe et al. (2003) | Q | 248, 272 |
| | 1.8 | | English and Carroll (2001) | Q | 230, 231 |
| | 3.1 | | Katritzky et al. (1998) | Q | |
| | $9.2\times10^{-1}$ | | Nirmalakhandan et al. (1997) | Q | |
| | 1.7 | | Russell et al. (1992) | Q | 279 |
| | 1.3 | | Suzuki et al. (1992) | Q | 232 |
| | 1.5 | | Duchowicz et al. (2020) | ? | 185, 21 |
| | 1.5 | | Mackay et al. (2006d) | ? | |
| | | 6400 | Kühne et al. (2005) | ? | |
| | 1.5 | | Abraham et al. (1990) | ? | |
| | | | Staudinger and Roberts (2001) | W | 560 |





Table A4.2: Heterocycles with nitrogen (C, H, N) (... continued)

| Substance Formula (Trivial Name) [CAS Registry Number] InChIKey | $H_s^{cp}$ (at $T^{\ominus}$) $\left[\dfrac{\text{mol}}{\text{m}^3\,\text{Pa}}\right]$ | $\dfrac{\mathrm{d}\ln H_s^{cp}}{\mathrm{d}(1/T)}$ [K] | Reference | Type | Note |
|---|---|---|---|---|---|
| 2,5-dimethylpyridine | 1.1 | 7000 | Andon et al. (1954) | M | 336 |
| $C_5H_3N(CH_3)_2$ | 1.5 | | Keshavarz et al. (2022) | Q | |
| (2,5-lutidine) | $2.7\times10^{-2}$ | | Duchowicz et al. (2020) | Q | 299 |
| [589-93-5] | $5.7\times10^{-1}$ | | Hilal et al. (2008) | Q | |
| XWKFPIODWVPXLX-UHFFFAOYSA-N | $2.1\times10^{-1}$ | | Modarresi et al. (2007) | Q | 67 |
| | | 6700 | Kühne et al. (2005) | Q | |
| | 1.2 | | Yaffe et al. (2003) | Q | 248, 249 |
| | 1.6 | | English and Carroll (2001) | Q | 230, 231 |
| | 2.9 | | Katritzky et al. (1998) | Q | |
| | $9.2\times10^{-1}$ | | Nirmalakhandan et al. (1997) | Q | |
| | 1.3 | | Suzuki et al. (1992) | Q | 232 |
| | 1.2 | | Meylan and Howard (1991) | Q | |
| | 1.1 | | Duchowicz et al. (2020) | ? | 185, 21 |
| | | 6900 | Kühne et al. (2005) | ? | |
| | 1.1 | | Abraham et al. (1990) | ? | |
| | | | Staudinger and Roberts (2001) | W | 560 |
| 2,6-dimethylpyridine | $9.2\times10^{-1}$ | 7300 | Brockbank (2013) | L | 1 |
| $C_5H_3N(CH_3)_2$ | $6.6\times10^{-1}$ | | Hawthorne et al. (1985) | M | |
| (2,6-lutidine) | $9.5\times10^{-1}$ | 7300 | Andon et al. (1954) | M | 336 |
| [108-48-5] | 1.5 | | Keshavarz et al. (2022) | Q | |
| OISVCGZHLKNMSJ-UHFFFAOYSA-N | $2.7\times10^{-2}$ | | Duchowicz et al. (2020) | Q | 299 |
| | $4.5\times10^{-1}$ | | Hilal et al. (2008) | Q | |
| | $6.8\times10^{-1}$ | | Modarresi et al. (2007) | Q | 67 |
| | | 6700 | Kühne et al. (2005) | Q | |
| | 1.6 | | English and Carroll (2001) | Q | 230, 231 |
| | 2.4 | | Katritzky et al. (1998) | Q | |
| | $9.5\times10^{-1}$ | | Nirmalakhandan et al. (1997) | Q | |
| | 1.3 | | Suzuki et al. (1992) | Q | 232 |
| | $9.5\times10^{-1}$ | | Duchowicz et al. (2020) | ? | 185, 21 |
| | $9.4\times10^{-1}$ | | Mackay et al. (2006d) | ? | |
| | | 6600 | Kühne et al. (2005) | ? | |
| | $9.5\times10^{-1}$ | | Abraham et al. (1990) | ? | |
| | | | Staudinger and Roberts (2001) | W | 560 |
| 3,4-dimethylpyridine | 2.7 | 6800 | Andon et al. (1954) | M | 336 |
| $C_5H_3N(CH_3)_2$ | 1.5 | | Keshavarz et al. (2022) | Q | |
| (3,4-lutidine) | $2.7\times10^{-2}$ | | Duchowicz et al. (2020) | Q | 299 |
| [583-58-4] | 1.3 | | Hilal et al. (2008) | Q | |
| NURQLCJSMXZBPC-UHFFFAOYSA-N | $3.8\times10^{-1}$ | | Modarresi et al. (2007) | Q | 67 |
| | | 6200 | Kühne et al. (2005) | Q | |
| | 2.7 | | Yaffe et al. (2003) | Q | 248, 249 |
| | 1.4 | | English and Carroll (2001) | Q | 230, 260 |
| | 2.8 | | Katritzky et al. (1998) | Q | |
| | $9.2\times10^{-1}$ | | Nirmalakhandan et al. (1997) | Q | |
| | 1.3 | | Suzuki et al. (1992) | Q | 232 |
| | 2.7 | | Duchowicz et al. (2020) | ? | 185, 21 |
| | | 6400 | Kühne et al. (2005) | ? | |



Table A4.2: Heterocycles with nitrogen (C, H, N) (. . . continued)

| Substance Formula (Trivial Name) [CAS Registry Number] InChIKey | $H_s^{cp}$ (at $T^{\ominus}$) $\left[\dfrac{\mathrm{mol}}{\mathrm{m^3\,Pa}}\right]$ | $\dfrac{\mathrm{d}\ln H_s^{cp}}{\mathrm{d}(1/T)}$ [K] | Reference | Type | Note |
|---|---|---|---|---|---|
| | 2.7 | | Abraham et al. (1990) | ? | |
| | | | Staudinger and Roberts (2001) | W | 560 |
| 3,5-dimethylpyridine $C_5H_3N(CH_3)_2$ (3,5-lutidine) [591-22-0] HWWYDZCSSYKIAD-UHFFFAOYSA-N | 1.4 | 6800 | Andon et al. (1954) | M | 336 |
| | 1.5 | | Keshavarz et al. (2022) | Q | |
| | $2.7\times10^{-2}$ | | Duchowicz et al. (2020) | Q | 184 |
| | $9.7\times10^{-1}$ | | Hilal et al. (2008) | Q | |
| | $5.9\times10^{-1}$ | | Modarresi et al. (2007) | Q | 67 |
| | | 6700 | Kühne et al. (2005) | Q | |
| | 1.5 | | Yaffe et al. (2003) | Q | 248, 249 |
| | 1.2 | | English and Carroll (2001) | Q | 230, 231 |
| | 3.3 | | Katritzky et al. (1998) | Q | |
| | $9.2\times10^{-1}$ | | Nirmalakhandan et al. (1997) | Q | |
| | 1.3 | | Suzuki et al. (1992) | Q | 232 |
| | 1.4 | | Duchowicz et al. (2020) | ? | 185, 21 |
| | | 6500 | Kühne et al. (2005) | ? | |
| | 1.4 | | Abraham et al. (1990) | ? | |
| | | | Staudinger and Roberts (2001) | W | 560 |
| 5-ethyl-2-methylpyridine $C_8H_{11}N$ [104-90-5] NTSLROIKFLNUIJ-UHFFFAOYSA-N | $5.2\times10^{-1}$ | | Duchowicz et al. (2020) | V | 186 |
| | $5.2\times10^{-1}$ | | HSDB (2015) | V | |
| | $2.7\times10^{-2}$ | | Duchowicz et al. (2020) | Q | |
| | $8.6\times10^{-1}$ | | Zhang et al. (2010) | Q | 287, 288 |
| | $3.8\times10^{-1}$ | | Zhang et al. (2010) | Q | 287, 289 |
| | $7.0\times10^{-1}$ | | Zhang et al. (2010) | Q | 287, 290 |
| | $6.2\times10^{-2}$ | | Zhang et al. (2010) | Q | 287, 291 |
| | $4.4\times10^{-1}$ | | Hilal et al. (2008) | Q | |
| | $1.4\times10^{-1}$ | | Modarresi et al. (2007) | Q | 67 |
| 2,4,6-trimethylpyridine $C_5H_2N(CH_3)_3$ (2,4,6-collidine) [108-75-8] BWZVCCNYKMEVEX-UHFFFAOYSA-N | 1.1 | | Duchowicz et al. (2020) | V | 186 |
| | 1.1 | | HSDB (2015) | V | |
| | $5.7\times10^{-2}$ | | Mackay et al. (2006d) | V | |
| | $5.7\times10^{-2}$ | | Mackay et al. (1995) | V | |
| | 1.1 | | Hilal et al. (2008) | C | |
| | $8.9\times10^{-3}$ | | Duchowicz et al. (2020) | Q | |
| | $5.4\times10^{-1}$ | | Hilal et al. (2008) | Q | |
| | 1.1 | | Modarresi et al. (2007) | Q | 67 |
| | | 7100 | Kühne et al. (2005) | Q | |
| | 2.5 | | Katritzky et al. (1998) | Q | |
| | | 8600 | Kühne et al. (2005) | ? | |
| indole $C_8H_7N$ [120-72-9] SIKJAQJRHWYJAI-UHFFFAOYSA-N | $1.9\times10^{1}$ | | Duchowicz et al. (2020) | V | 186 |
| | $1.9\times10^{1}$ | | HSDB (2015) | V | |
| | 7.1 | | Mackay et al. (2006d) | V | |
| | 7.1 | | Mackay et al. (1995) | V | |
| | $1.6\times10^{1}$ | | Yaws (2003) | X | 258 |
| | $1.6\times10^{1}$ | | Yaws (2003) | X | 237, 12 |
| | $1.5\times10^{1}$ | | Howard and Meylan (1997) | X | 446 |
| | 5.3 | | Dupeux et al. (2022) | Q | 259 |



Table A4.2: Heterocycles with nitrogen (C, H, N) (... continued)

| Substance Formula (Trivial Name) [CAS Registry Number] InChIKey | $H_s^{cp}$ (at $T^{\ominus}$) $\left[\dfrac{\text{mol}}{\text{m}^3\,\text{Pa}}\right]$ | $\dfrac{\text{d}\ln H_s^{cp}}{\text{d}(1/T)}$ [K] | Reference | Type | Note |
|---|---|---|---|---|---|
| | $1.6\times10^1$ | | Duchowicz et al. (2020) | Q | |
| | $7.0\times10^1$ | | Gharagheizi et al. (2012) | Q | |
| | $1.7\times10^1$ | | Gharagheizi et al. (2010) | Q | 246 |
| | 9.0 | | Hilal et al. (2008) | Q | |
| | 9.9 | | Yaws (1999) | ? | 21, 12 |
| 2-ethyl-3,5-dimethylpyrazine $C_8H_{12}N_2$ [13925-07-0] JZBCTZLGKSYRSF-UHFFFAOYSA-N | 2.9 | 8500 | Wieland et al. (2015) | M | 561 |
| 2-isobutylpyrazine $C_4N_2H_3C_4H_9$ [29460-92-2] YAIMUUJMEBJXAA-UHFFFAOYSA-N | 2.0 $1.2\times10^1$ 1.4 | | Buttery et al. (1971) Keshavarz et al. (2022) Nirmalakhandan et al. (1997) | M Q Q | |
| 2-(1-methylpropyl)-pyrazine $C_8H_{12}N_2$ [29460-93-3] NFFQZEXYZVZKNN-UHFFFAOYSA-N | 1.6 2.2 | | Hilal et al. (2008) Modarresi et al. (2007) | Q Q | 67 |
| 5-ethenyl-2-methylpyridine $C_8H_9N$ [140-76-1] VJOWMORERYNYON-UHFFFAOYSA-N | 2.2 | | HSDB (2015) | Q | 99 |
| 4-(1,1-dimethylethyl)-pyridine $C_9H_{13}N$ (4-*tert*-butylpyridine) [3978-81-2] YSHMQTRICHYLGF-UHFFFAOYSA-N | $3.9\times10^{-1}$ $7.5\times10^{-1}$ $7.5\times10^{-1}$ | 7000 | Hilal et al. (2008) Abraham et al. (1990) Arnett and Chawla (1979) | Q ? ? | 559 |
| nornicotine $C_9H_{12}N_2$ [494-97-3] MYKUKUCHPMASKF-UHFFFAOYSA-N | $7.2\times10^3$ | | HSDB (2015) | Q | 447 |
| 2,4-diamino-6-phenyl-1,3,5-triazine $C_9H_9N_5$ [91-76-9] GZVHEAJQGPRDLQ-UHFFFAOYSA-N | $2.4\times10^5$ | | HSDB (2015) | Q | 99 |
| N-methylindole $C_9H_9N$ [603-76-9] BLRHMMGNCXNXJL-UHFFFAOYSA-N | 8.7 | | Ebert et al. (2023) | ? | 365 |



Table A4.2: Heterocycles with nitrogen (C, H, N) (...continued)

| Substance Formula (Trivial Name) [CAS Registry Number] InChIKey | $H_s^{cp}$ (at $T^{\ominus}$) $\left[\dfrac{\mathrm{mol}}{\mathrm{m^3\,Pa}}\right]$ | $\dfrac{\mathrm{d}\ln H_s^{cp}}{\mathrm{d}(1/T)}$ [K] | Reference | Type | Note |
|---|---|---|---|---|---|
| 3-methylindole $C_9H_9N$ (skatole) [83-34-1] ZFRKQXVRDFCRJG-UHFFFAOYSA-N | $2.9\times10^{-5}$ 4.7 | -2500 | Wetlaufer et al. (1964) HSDB (2015) | M V | |
| 2,3-diethyl-5-methylpyrazine $C_9H_{14}N_2$ [18138-04-0] PSINWXIDJYEXLO-UHFFFAOYSA-N | $8.1\times10^{-1}$ | | Roberts and Pollien (1997) | M | |
| benzo[$b$]pyridine $C_9H_7N$ (quinoline) [91-22-5] SMWDFEZZVXVKRB-UHFFFAOYSA-N | 5.9 5.8 $3.8\times10^1$ $3.8\times10^1$ 6.0 $3.9\times10^1$ 6.4 $3.7\times10^1$ $9.3\times10^{-1}$ $5.7\times10^1$ 6.4 5.8 6.1 9.0 $3.2\times10^{-1}$ $3.4\times10^1$ $1.4\times10^1$ 6.5 | 5400 7300 7300 | Duchowicz et al. (2020) HSDB (2015) Mackay et al. (2006d) Mackay et al. (1995) Meylan and Howard (1991) Smith and Bomberger (1980) Abraham et al. (1994a) Goldstein (1982) Duchowicz et al. (2020) Gharagheizi et al. (2012) Hilal et al. (2008) Modarresi et al. (2007) Kühne et al. (2005) Yaffe et al. (2003) English and Carroll (2001) Katritzky et al. (1998) Nirmalakhandan et al. (1997) Meylan and Howard (1991) Kühne et al. (2005) Yaws (1999) | V V V V V V R X Q Q Q Q Q Q Q Q Q Q ? ? | 186 24 298 67 248, 249 230, 231 21, 12 |
| benzo[$c$]pyridine $C_9H_7N$ (isoquinoline) [119-65-3] AWJUIBRHMBBTKR-UHFFFAOYSA-N | $5.2\times10^{-2}$ $5.2\times10^{-2}$ 5.7 5.6 9.2 3.8 | | Mackay et al. (2006d) Mackay et al. (1995) Yaws (2003) Gharagheizi et al. (2010) Hilal et al. (2008) Yaws (1999) | V V X Q Q ? | 237, 12 246 21, 12 |
| nicotine $C_{10}H_{14}N_2$ [54-11-5] SNICXCGAKADSCV-SNVBAGLBSA-N | $3.3\times10^3$ | | HSDB (2015) | Q | 99 |
| 2,2'-bipyridine $C_{10}H_8N_2$ [366-18-7] ROFVEXUMMXZLPA-UHFFFAOYSA-N | $1.8\times10^4$ | | HSDB (2015) | Q | 545 |



Table A4.2: Heterocycles with nitrogen (C, H, N) (...continued)

| Substance Formula (Trivial Name) [CAS Registry Number] InChIKey | $H_s^{cp}$ (at $T^\ominus$) $\left[\dfrac{\mathrm{mol}}{\mathrm{m^3\,Pa}}\right]$ | $\dfrac{\mathrm{d}\ln H_s^{cp}}{\mathrm{d}(1/T)}$ [K] | Reference | Type | Note |
|---|---|---|---|---|---|
| 4,4'-bipyridine $C_{10}H_8N_2$ [553-26-4] MWVTWFVJZLCBMC-UHFFFAOYSA-N | $3.9\times10^3$ $1.7\times10^1$ | | Duchowicz et al. (2020) Duchowicz et al. (2020) | V Q | 186 |
| 2-methylquinoline $C_{10}H_9N$ [91-63-4] SMUQFGGVLNAIOZ-UHFFFAOYSA-N | $1.4\times10^1$ | | Ebert et al. (2023) | ? | 316 |
| 4-methylquinoline $C_{10}H_9N$ [491-35-0] MUDSDYNRBDKLGK-UHFFFAOYSA-N | $1.3\times10^1$ | | HSDB (2015) | Q | 99 |
| MEIQX $C_{11}H_{11}N_5$ (2-amino-3,8-dimethylimidazo[4,5-f]quinoxaline) [77500-04-0] DVCCCQNKIYNAKB-UHFFFAOYSA-N | $6.2\times10^7$ | | HSDB (2015) | Q | 99 |
| 3-(phenylazo)-2,6-pyridinediamine $C_{11}H_{11}N_5$ (phenazopyridine) [94-78-0] QPFYXYFORQJZEC-UHFFFAOYSA-N | $3.0\times10^9$ | | HSDB (2015) | Q | 99 |
| 2-amino-9H-pyrido[2,3-b]indole $C_{11}H_9N_3$ [26148-68-5] FJTNLJLPLJDTRM-UHFFFAOYSA-N | $2.5\times10^8$ | | HSDB (2015) | Q | 99 |
| carbazole $C_{12}H_9N$ [86-74-8] UJOBWOGCFQCDNV-UHFFFAOYSA-N | $9.4\times10^1$ $9.3\times10^1$ $6.6\times10^{-2}$ $6.6\times10^{-2}$ $6.3\times10^{-2}$ $2.0\times10^1$ $6.3\times10^1$ $1.1\times10^2$ $8.5\times10^1$ | 4300 4300 | Brockbank (2013) Odabasi et al. (2006) Mackay et al. (2006d) Mackay et al. (1995) Smith and Bomberger (1980) Keshavarz et al. (2022) Duchowicz et al. (2020) HSDB (2015) Duchowicz et al. (2020) | L M V V V Q Q Q ? | 1 24 99 185, 21 |
| o-phenanthroline $C_{12}H_8N_2$ [66-71-7] DGEZNRSVGBDHLK-UHFFFAOYSA-N | $1.1\times10^5$ $9.9\times10^3$ $1.2\times10^5$ $4.4\times10^2$ | | Zhang et al. (2010) Zhang et al. (2010) Zhang et al. (2010) Zhang et al. (2010) | Q Q Q Q | 287, 288 287, 289 287, 290 287, 291 |



Table A4.2: Heterocycles with nitrogen (C, H, N) (. . . continued)

| Substance Formula (Trivial Name) [CAS Registry Number] InChIKey | $H_s^{cp}$ (at $T^\ominus$) $\left[\dfrac{\mathrm{mol}}{\mathrm{m^3\,Pa}}\right]$ | $\dfrac{\mathrm{d\ln}H_s^{cp}}{\mathrm{d}(1/T)}$ [K] | Reference | Type | Note |
|---|---|---|---|---|---|
| benzyladenine $C_{12}H_{11}N_5$ [1214-39-7] NWBJYWHLCVSVIJ-UHFFFAOYSA-N | $1.1\times10^8$ $1.1\times10^{-1}$ | | HSDB (2015) Maniere et al. (2011) | V ? | 241, 165 |
| diquat $C_{12}H_{12}N_2$ [2764-72-9] SYJFEGQWDCRVNX-UHFFFAOYSA-N | $2.8\times10^5$ | | Ebert et al. (2023) | ? | 318 |
| MEIQ $C_{12}H_{12}N_4$ (2-amino-3,4-dimethylimidazo[4,5-f]quinoxaline) [77094-11-2] GMGWMIJIGUYNAY-UHFFFAOYSA-N | $2.5\times10^7$ | | HSDB (2015) | Q | 99 |
| pyrimethanil $C_{12}H_{13}N_3$ [53112-28-0] ZLIBICFPKPWGIZ-UHFFFAOYSA-N | $3.4\times10^2$ $2.8\times10^2$ $2.8\times10^2$ $9.2\times10^2$ $3.9$ $2.8\times10^2$ | 15000 | Feigenbrugel and Le Calvé (2021) Duchowicz et al. (2020) Matthews (1998) Duchowicz et al. (2020) HSDB (2015) Maniere et al. (2011) | M V X Q Q ? | 186 562, 12 99 241, 165 |
| paraquat $C_{12}H_{14}N_2$ [4685-14-7] INFDPOAKFNIJBF-UHFFFAOYSA-N | $>2.4\times10^8$ | | HSDB (2015) | V | |
| N,N-dimethyltryptamine $C_{12}H_{16}N_2$ [61-50-7] DMULVCHRPCFFGV-UHFFFAOYSA-N | $1.5\times10^4$ | | HSDB (2015) | Q | 99 |
| benzo[$f$]quinoline $C_{13}H_9N$ [85-02-9] HCAUQPZEWLULFJ-UHFFFAOYSA-N | $5.7\times10^1$ $1.0\times10^2$ $1.0\times10^2$ $3.6$ | | Duchowicz et al. (2020) Mackay et al. (2006d) Mackay et al. (1995) Smith and Bomberger (1980) Duchowicz et al. (2020) | V V V V Q | 186 558 24 |
| 2,6-bis-(1,1-dimethylethyl)-pyridine $C_{13}H_{21}N$ (2,6-di-$tert$-butylpyridine) [585-48-8] UWKQJZCTQGMHKD-UHFFFAOYSA-N | $8.0\times10^{-4}$ $2.8\times10^{-1}$ | 6900 | Arnett and Chawla (1979) Arnett and Chawla (1979) | M V | 559 563 |



Table A4.2: Heterocycles with nitrogen (C, H, N) (...continued)

| Substance<br>Formula<br>(Trivial Name)<br>[CAS Registry Number]<br>InChIKey | $H_s^{cp}$ (at $T^\ominus$) $\left[\dfrac{\text{mol}}{\text{m}^3\,\text{Pa}}\right]$ | $\dfrac{\text{d}\ln H_s^{cp}}{\text{d}(1/T)}$ [K] | Reference | Type | Note |
|---|---|---|---|---|---|
| PHIP<br>$C_{13}H_{12}N_4$<br>(2-amino-1-methyl-6-phenylimidazo[4,5-b]pyridine)<br>[105650-23-5]<br>UQVKZNNCIHJZLS-UHFFFAOYSA-N | $3.5\times10^7$ | | HSDB (2015) | Q | 99 |
| N,N'-diphenylguanidine<br>$C_{13}H_{13}N_3$<br>[102-06-7]<br>OWRCNXZUPFZXOS-UHFFFAOYSA-N | $1.4\times10^6$ | | HSDB (2015) | Q | 99 |
| acridine<br>$C_{13}H_9N$<br>[260-94-6]<br>DZBUGLKDJFMEHC-UHFFFAOYSA-N | $3.3\times10^1$<br>$3.3\times10^1$<br>$2.5\times10^1$ | | Mackay et al. (2006d)<br>Mackay et al. (1995)<br>HSDB (2015) | V<br>V<br>Q | <br><br>99 |
| phenanthridine<br>$C_{13}H_9N$<br>[229-87-8]<br>RDOWQLZANAYVLL-UHFFFAOYSA-N | $6.0\times10^2$<br>3.6 | | Duchowicz et al. (2020)<br>Duchowicz et al. (2020) | V<br>Q | 186 |
| mepanipyrim<br>$C_{14}H_{13}N_3$<br>[110235-47-7]<br>CIFWZNRJIBNXRE-UHFFFAOYSA-N | $6.0\times10^2$ | | Maniere et al. (2011) | ? | 241, 165 |
| cyprodinil<br>$C_{14}H_{15}N_3$<br>[121552-61-2]<br>HAORKNGNJCEJBX-UHFFFAOYSA-N | $1.2\times10^2$<br>$1.2\times10^2$<br>$2.9\times10^3$ | | Duchowicz et al. (2020)<br>HSDB (2015)<br>Duchowicz et al. (2020) | V<br>V<br>Q | 186 |
| imiquimod<br>$C_{14}H_{16}N_4$<br>[99011-02-6]<br>DOUYETYNHWVLEO-UHFFFAOYSA-N | $1.2\times10^7$ | | HSDB (2015) | Q | 99 |
| ametoctradin<br>$C_{15}H_{25}N_5$<br>[865318-97-4]<br>GGKQIOFASHYUJZ-UHFFFAOYSA-N | $2.4\times10^6$ | | Maniere et al. (2011) | ? | 12, 165 |
| ferimzone<br>$C_{15}H_{18}N_4$<br>[89269-64-7]<br>GOWLARCWZRESHU-UHFFFAOYSA-N | $1.6\times10^5$<br>$2.4\times10^2$ | | Duchowicz et al. (2020)<br>Duchowicz et al. (2020) | V<br>Q | 186 |
| benz[c]acridine<br>$C_{17}H_{11}N$<br>[225-51-4]<br>OAPPEBNXKAKQGS-UHFFFAOYSA-N | $3.7\times10^2$ | | HSDB (2015) | Q | 447 |



Table A4.2: Heterocycles with nitrogen (C, H, N) (...continued)

| Substance<br>Formula<br>(Trivial Name)<br>[CAS Registry Number]<br>InChIKey | $H_s^{cp}$<br>(at $T^{\ominus}$)<br>$\left[\dfrac{\mathrm{mol}}{\mathrm{m^3\,Pa}}\right]$ | $\dfrac{\mathrm{d}\ln H_s^{cp}}{\mathrm{d}(1/T)}$<br><br>[K] | Reference | Type | Note |
|---|---|---|---|---|---|
| 6-pentyl-1,2,3,4,7,8,9,10-<br>octahydrophenanthridine | $4.5\times10^{-1}$ | | Zhang et al. (2010) | Q | 287, 288 |
| $C_{18}H_{27}N$ | $2.0\times10^{1}$ | | Zhang et al. (2010) | Q | 287, 289 |
| [10594-03-3] | 6.2 | | Zhang et al. (2010) | Q | 287, 290 |
| FNUATPIDZQSFPD-UHFFFAOYSA-N | $2.2\times10^{-2}$ | | Zhang et al. (2010) | Q | 287, 291 |
| 2-diphenylmethylpiperidine | $6.6\times10^{1}$ | | HSDB (2015) | Q | 99 |
| $C_{18}H_{21}N$ | | | | | |
| (desoxypipradrol) | | | | | |
| [519-74-4] | | | | | |
| RWTNXJXZVGHMGI-UHFFFAOYSA-N | | | | | |
| fenpropidin | $7.1\times10^{2}$ | 6200 | Feigenbrugel and Le Calvé (2021) | M | 33 |
| $C_{19}H_{31}N$ | $1.1\times10^{2}$ | | Duchowicz et al. (2020) | V | 186 |
| [67306-00-7] | $9.9\times10^{-2}$ | | Matthews (1998) | X | 562, 12 |
| MGNFYQILYYYUBS-UHFFFAOYSA-N | $9.5\times10^{-2}$ | | Duchowicz et al. (2020) | Q | |
| | $9.3\times10^{-2}$ | | Maniere et al. (2011) | ? | 165 |
| N,N-bis(2-ethylhexyl)-1H-1,2,4-<br>triazole-1-methanamine | $2.9\times10^{1}$ | | Ebert et al. (2023) | ? | 365 |
| $C_{19}H_{38}N_4$ | | | | | |
| [91273-04-0] | | | | | |
| AVBBHCMDRGQBNW-UHFFFAOYSA-N | | | | | |
| 7H-dibenzo[$c,g$]carbazole | $2.1\times10^{3}$ | | Smith and Bomberger (1980) | V | 24 |
| $C_{20}H_{13}N$ | $4.0\times10^{3}$ | | HSDB (2015) | Q | 99 |
| [194-59-2] | | | | | |
| STJXCDGCXVZHDU-UHFFFAOYSA-N | | | | | |
| porphyrin | $3.9\times10^{5}$ | | Abraham et al. (2019) | Q | |
| $C_{20}H_{14}N_4$ | | | | | |
| (porphin) | | | | | |
| [101-60-0] | | | | | |
| RKCAIXNGYQCCAL-YYOYBPFYSA-N | | | | | |
| dibenz[$a,j$]acridine | $5.2\times10^{3}$ | | HSDB (2015) | Q | 99 |
| $C_{21}H_{13}N$ | | | | | |
| [224-42-0] | | | | | |
| ANUCHZVCBDOPOX-UHFFFAOYSA-N | | | | | |
| dibenz[$a,h$]acridine | $5.2\times10^{3}$ | | HSDB (2015) | Q | 99 |
| $C_{21}H_{13}N$ | | | | | |
| [226-36-8] | | | | | |
| JNCSIWAONQTVCF-UHFFFAOYSA-N | | | | | |
| 1,3,5-tricyclohexylhexahydro-<br>1,3,5-triazine | $1.7\times10^{-2}$ | | Zhang et al. (2010) | Q | 287, 288 |
| $C_{21}H_{39}N_3$ | $4.0\times10^{5}$ | | Zhang et al. (2010) | Q | 287, 289 |
| [6281-14-7] | $1.2\times10^{5}$ | | Zhang et al. (2010) | Q | 287, 290 |
| ZLLRUEJANKJPQE-UHFFFAOYSA-N | $1.5\times10^{4}$ | | Zhang et al. (2010) | Q | 287, 291 |



### A4.3 Nitriles (C, H, N)

Table A4.3: Nitriles (C, H, N)

| Substance Formula (Trivial Name) [CAS Registry Number] InChIKey | $H_s^{cp}$ (at $T^\ominus$) $\left[\dfrac{\mathrm{mol}}{\mathrm{m^3\,Pa}}\right]$ | $\dfrac{\mathrm{d}\ln H_s^{cp}}{\mathrm{d}(1/T)}$ [K] | Reference | Type | Note |
|---|---|---|---|---|---|
| cyano radical CN [2074-87-5] JEVCWSUVFOYBFI-UHFFFAOYSA-N | $7.8\times10^{-4}$ | 1400 | Berdnikov and Bazhin (1970) | T | 47 |
| hydrogen cyanide HCN (hydrocyanic acid) [74-90-8] LELOWRISYMNNSU-UHFFFAOYSA-N | $8.9\times10^{-2}$ | 8200 | Burkholder et al. (2019) | L | |
| | $8.9\times10^{-2}$ | 8200 | Burkholder et al. (2015) | L | |
| | $1.7\times10^{-1}$ | 4400 | Yoo et al. (1986) | L | 1 |
| | $1.1\times10^{-1}$ | 5000 | Edwards et al. (1978) | L | 1 |
| | $8.9\times10^{-2}$ | 8200 | Ma et al. (2010a) | M | |
| | $7.5\times10^{-2}$ | | Riveros et al. (1998) | M | 12 |
| | $1.2\times10^{-1}$ | | Fredenhagen and Wellmann (1932b) | M | |
| | $9.2\times10^{-2}$ | | Hine and Weimar (1965) | R | |
| | $9.9\times10^{-2}$ | 4200 | Edwards et al. (1975) | T | 1 |
| | $7.6\times10^{-2}$ | 2900 | Kotlik and Lebedeva (1974) | X | 564 |
| | $7.4\times10^{-2}$ | | Gaffney and Senum (1984) | X | 389, 491 |
| | $5.0\times10^{-2}$ | | Keshavarz et al. (2022) | Q | |
| | $4.2\times10^{-2}$ | | Duchowicz et al. (2020) | Q | 184 |
| | $3.9\times10^{-2}$ | | Hilal et al. (2008) | Q | |
| | $1.6\times10^{-1}$ | | Modarresi et al. (2007) | Q | 67 |
| | 4.7 | | Katritzky et al. (1998) | Q | |
| | $7.4\times10^{-2}$ | | Duchowicz et al. (2020) | ? | 185, 21 |
| | $1.1\times10^{-1}$ | | Yaws (1999) | ? | 21 |
| ethanenitrile CH₃CN (acetonitrile) [75-05-8] WEVYAHXRMPXWCK-UHFFFAOYSA-N | $5.2\times10^{-1}$ | 4000 | Burkholder et al. (2019) | L | |
| | $5.2\times10^{-1}$ | 4000 | Burkholder et al. (2015) | L | |
| | $4.9\times10^{-1}$ | 4300 | Brockbank (2013) | L | 1 |
| | $5.2\times10^{-1}$ | 4000 | Sander et al. (2011) | L | |
| | $5.2\times10^{-1}$ | 4000 | Sander et al. (2006) | L | |
| | $3.8\times10^{-1}$ | 4200 | Plyasunov et al. (2006) | L | |
| | $5.0\times10^{-1}$ | 4100 | Fogg and Sangster (2003) | L | |
| | $5.1\times10^{-1}$ | 4000 | Staudinger and Roberts (2001) | L | |
| | $4.7\times10^{-1}$ | 3500 | Arijs and Brasseur (1986) | L | |
| | $6.0\times10^{-1}$ | 6300 | Hiatt (2013) | M | |
| | $5.2\times10^{-1}$ | 4000 | Ji and Evans (2007) | M | |
| | $4.9\times10^{-1}$ | | Bebahani et al. (2002) | M | |
| | $3.2\times10^{-1}$ | 3300 | Hovorka et al. (2002) | M | 11 |
| | $3.4\times10^{-1}$ | | Welke et al. (1998) | M | |
| | $5.3\times10^{-1}$ | 4100 | Benkelberg et al. (1995) | M | |
| | $4.6\times10^{-1}$ | | Li and Carr (1993) | M | |
| | $8.2\times10^{-2}$ | | Yu (1992) | M | 12 |
| | $4.8\times10^{-1}$ | 3900 | Snider and Dawson (1985) | M | |
| | $5.3\times10^{-1}$ | 4100 | Hamm et al. (1984) | M | |
| | $3.7\times10^{-2}$ | | Abraham and Acree (2007) | V | |
| | $5.0\times10^{-1}$ | | Hwang et al. (1992) | V | |
| | $2.9\times10^{-1}$ | | Hine and Weimar (1965) | R | |





Table A4.3: Nitriles (C, H, N) (...continued)

| Substance / Formula / (Trivial Name) / [CAS Registry Number] / InChIKey | $H_s^{cp}$ (at $T^\ominus$) $\left[\dfrac{\mathrm{mol}}{\mathrm{m}^3\,\mathrm{Pa}}\right]$ | $\dfrac{\mathrm{d}\ln H_s^{cp}}{\mathrm{d}(1/T)}$ [K] | Reference | Type | Note |
|---|---|---|---|---|---|
| | $2.9\times10^{-1}$ | | Gaffney and Senum (1984) | X | 389 |
| | $4.7\times10^{-1}$ | | Hayer et al. (2022) | Q | 20 |
| | $3.4\times10^{-1}$ | | Keshavarz et al. (2022) | Q | |
| | $1.0\times10^{-1}$ | | Duchowicz et al. (2020) | Q | |
| | $2.9\times10^{-1}$ | | Li et al. (2014) | Q | 241 |
| | $7.7\times10^{-1}$ | | Hilal et al. (2008) | Q | |
| | $3.7\times10^{-1}$ | | Modarresi et al. (2007) | Q | 67 |
| | | 4200 | Kühne et al. (2005) | Q | |
| | $2.9\times10^{-1}$ | | Yaffe et al. (2003) | Q | 248, 249 |
| | $2.9\times10^{-1}$ | | English and Carroll (2001) | Q | 230, 231 |
| | $4.4$ | | Katritzky et al. (1998) | Q | |
| | $2.9\times10^{-2}$ | | Nirmalakhandan et al. (1997) | Q | |
| | $3.1\times10^{-1}$ | | Suzuki et al. (1992) | Q | 232 |
| | $2.9\times10^{-1}$ | | Duchowicz et al. (2020) | ? | 185, 21 |
| | $3.6\times10^{-1}$ | | Mackay et al. (2006d) | ? | |
| | | 4300 | Kühne et al. (2005) | ? | |
| | $4.9\times10^{-1}$ | | Yaws (1999) | ? | 21 |
| | $4.9\times10^{-1}$ | | Yaws and Yang (1992) | ? | 21 |
| | $2.9\times10^{-1}$ | | Abraham et al. (1990) | ? | |
| propanenitrile $C_2H_5CN$ (propionitrile) [107-12-0] FVSKHRXBFJPNKK-UHFFFAOYSA-N | $3.4\times10^{-1}$ | 4500 | Brockbank (2013) | L | 1 |
| | $2.4\times10^{-1}$ | 4800 | Plyasunov et al. (2006) | L | |
| | $4.3\times10^{-1}$ | 6200 | Hiatt (2013) | M | |
| | $3.3\times10^{-1}$ | 4600 | Ji and Evans (2007) | M | |
| | $1.8\times10^{-1}$ | | Hovorka et al. (2002) | M | 38 |
| | $2.5\times10^{-1}$ | | Li and Carr (1993) | M | |
| | $1.9\times10^{-1}$ | | Hawthorne et al. (1985) | M | |
| | $2.6\times10^{-1}$ | | Butler and Ramchandani (1935) | M | |
| | $3.1\times10^{-1}$ | | Mackay et al. (2006d) | V | |
| | $3.1\times10^{-1}$ | | Mackay et al. (1995) | V | |
| | $1.7\times10^{-1}$ | | Howard (1990) | X | 412 |
| | $3.5\times10^{-1}$ | | Keshavarz et al. (2022) | Q | |
| | $1.0\times10^{-1}$ | | Duchowicz et al. (2020) | Q | |
| | $5.0\times10^{-1}$ | | Hilal et al. (2008) | Q | |
| | $2.6\times10^{-1}$ | | Modarresi et al. (2007) | Q | 67 |
| | $2.9\times10^{-1}$ | | English and Carroll (2001) | Q | 230, 231 |
| | $5.6\times10^{-1}$ | | Russell et al. (1992) | Q | 279 |
| | $2.4\times10^{-1}$ | | Suzuki et al. (1992) | Q | 232 |
| | $2.7\times10^{-1}$ | | Duchowicz et al. (2020) | ? | 185, 21 |
| | $2.6\times10^{-1}$ | | Mackay et al. (2006d) | ? | |
| | $2.9\times10^{-1}$ | | Yaws (1999) | ? | 21 |
| | $2.7\times10^{-1}$ | | Abraham et al. (1990) | ? | |





Table A4.3: Nitriles (C, H, N) (...continued)

| Substance Formula (Trivial Name) [CAS Registry Number] InChIKey | $H_s^{cp}$ (at $T^{\ominus}$) $\left[\dfrac{\mathrm{mol}}{\mathrm{m^3\,Pa}}\right]$ | $\dfrac{\mathrm{d}\ln H_s^{cp}}{\mathrm{d}(1/T)}$ [K] | Reference | Type | Note |
|---|---|---|---|---|---|
| butanenitrile | $1.8\times10^{-1}$ | 5100 | Brockbank (2013) | L | 1 |
| $C_3H_7CN$ | $1.8\times10^{-1}$ | 5100 | Plyasunov et al. (2006) | L | |
| (butyronitrile) | $2.7\times10^{-1}$ | 5100 | Ji and Evans (2007) | M | |
| [109-74-0] | $1.3\times10^{-1}$ | | Ramachandran et al. (1996) | M | |
| KVNRLNFWIYMESJ-UHFFFAOYSA-N | $1.9\times10^{-1}$ | | Li and Carr (1993) | M | |
| | $1.4\times10^{-1}$ | | Hawthorne et al. (1985) | M | |
| | $1.9\times10^{-1}$ | | Butler and Ramchandani (1935) | M | |
| | $1.8\times10^{-1}$ | | Yaws (2003) | X | 258 |
| | $1.8\times10^{-1}$ | | Yaws (2003) | X | 237 |
| | $3.7\times10^{-1}$ | | Dupeux et al. (2022) | Q | 259 |
| | $1.2\times10^{-1}$ | | Keshavarz et al. (2022) | Q | |
| | $1.0\times10^{-1}$ | | Duchowicz et al. (2020) | Q | 184 |
| | $1.8\times10^{-1}$ | | Gharagheizi et al. (2010) | Q | 246 |
| | $3.5\times10^{-1}$ | | Hilal et al. (2008) | Q | |
| | $1.5\times10^{-1}$ | | Modarresi et al. (2007) | Q | 67 |
| | | 4900 | Kühne et al. (2005) | Q | |
| | $1.5\times10^{-1}$ | | Yaffe et al. (2003) | Q | 248, 249 |
| | $5.7\times10^{-2}$ | | English and Carroll (2001) | Q | 230, 274 |
| | $1.9\times10^{-2}$ | | Nirmalakhandan et al. (1997) | Q | |
| | $5.4\times10^{-1}$ | | Russell et al. (1992) | Q | 279 |
| | $1.9\times10^{-1}$ | | Suzuki et al. (1992) | Q | 232 |
| | $1.9\times10^{-1}$ | | Duchowicz et al. (2020) | ? | 185, 21 |
| | $1.9\times10^{-1}$ | | Mackay et al. (2006d) | ? | |
| | | 4700 | Kühne et al. (2005) | ? | |
| | $1.8\times10^{-1}$ | | Yaws (1999) | ? | 21 |
| | $1.9\times10^{-1}$ | | Abraham et al. (1990) | ? | |
| 2-methylpropanenitrile | $1.1\times10^{-1}$ | 5200 | Brockbank (2013) | L | 1 |
| $C_4H_7N$ | $1.0\times10^{-1}$ | 4800 | Plyasunov et al. (2006) | L | |
| (isobutyronitrile) | $9.4\times10^{-2}$ | | Li and Carr (1993) | M | |
| [78-82-0] | $1.8\times10^{-1}$ | | HSDB (2015) | Q | 99 |
| LRDFRRGEGBBSRN-UHFFFAOYSA-N | $1.9\times10^{-1}$ | | Hilal et al. (2008) | Q | |
| | | 4900 | Kühne et al. (2005) | Q | |
| | | 5100 | Kühne et al. (2005) | ? | |
| pentanenitrile | $1.6\times10^{-1}$ | 6100 | Brockbank (2013) | L | 1 |
| $C_4H_9CN$ | $1.4\times10^{-1}$ | 5500 | Plyasunov et al. (2006) | L | |
| (butyl cyanide; valeronitrile) | $1.4\times10^{-1}$ | | Li and Carr (1993) | M | |
| [110-59-8] | $1.6\times10^{-1}$ | | Amoore and Buttery (1978) | V | |
| RFFFKMOABOFIDF-UHFFFAOYSA-N | $2.7\times10^{-1}$ | | Hilal et al. (2008) | Q | |
| | $1.3\times10^{-1}$ | | Modarresi et al. (2007) | Q | 67 |
| | $1.5\times10^{-1}$ | | Yaffe et al. (2003) | Q | 248, 272 |
| | $4.1\times10^{-2}$ | | English and Carroll (2001) | Q | 230, 231 |
| | $1.5\times10^{-2}$ | | Nirmalakhandan et al. (1997) | Q | |
| | $1.5\times10^{-1}$ | | Abraham et al. (1990) | ? | |





Table A4.3: Nitriles (C, H, N) (...continued)

| Substance Formula (Trivial Name) [CAS Registry Number] InChIKey | $H_s^{cp}$ (at $T^\ominus$) $\left[\dfrac{\mathrm{mol}}{\mathrm{m^3\,Pa}}\right]$ | $\dfrac{\mathrm{d}\ln H_s^{cp}}{\mathrm{d}(1/T)}$ [K] | Reference | Type | Note |
|---|---|---|---|---|---|
| hexanenitrile $C_6H_{11}N$ [628-73-9] AILKHAQXUAOOFU-UHFFFAOYSA-N | $1.2\times10^{-1}$ $2.3\times10^{-1}$ | | Plyasunov et al. (2006) Hilal et al. (2008) | L Q | |
| heptanenitrile $C_7H_{13}N$ [629-08-3] SDAXRHHPNYTELL-UHFFFAOYSA-N | $1.6\times10^{-1}$ | | Hilal et al. (2008) | Q | |
| octanenitrile $C_8H_{15}N$ [124-12-9] YSIMAPNUZAVQER-UHFFFAOYSA-N | $5.7\times10^{-2}$ $1.3\times10^{-1}$ | | Plyasunov et al. (2006) Hilal et al. (2008) | L Q | |
| nonanenitrile $C_9H_{17}N$ [2243-27-8] PLZZPPHAMDJOSR-UHFFFAOYSA-N | $1.0\times10^{-1}$ | | Hilal et al. (2008) | Q | |
| decanenitrile $C_{10}H_{19}N$ [1975-78-6] HBZDPWBWBJMYRY-UHFFFAOYSA-N | $8.0\times10^{-2}$ | | Hilal et al. (2008) | Q | |
| undecanenitrile $C_{11}H_{21}N$ [2244-07-7] SZKKNEOUHLFYNA-UHFFFAOYSA-N | $6.1\times10^{-2}$ | | Hilal et al. (2008) | Q | |
| cyclohexanecarbonitrile $C_7H_{11}N$ [766-05-2] VBWIZSYFQSOUFQ-UHFFFAOYSA-N | $7.3\times10^{-1}$ | | Hilal et al. (2008) | Q | |
| ethanedinitrile $C_2N_2$ (cyanogen) [460-19-5] JMANVNJQNLATNU-UHFFFAOYSA-N | $1.8\times10^{-3}$ $1.8\times10^{-3}$ $1.9\times10^{-3}$ $1.9\times10^{1}$ $1.8\times10^{-3}$ $2.6\times10^{-3}$ $2.9\times10^{-2}$ $1.9\times10^{-3}$ $1.8\times10^{-3}$ | | Duchowicz et al. (2020) HSDB (2015) Yaws (2003) Duchowicz et al. (2020) Gharagheizi et al. (2010) Hilal et al. (2008) Modarresi et al. (2007) Yaws (1999) Yaws and Yang (1992) | V V X Q Q Q Q ? ? | 186  237, 12  246  67 21, 12 21, 12 |
| pentanedinitrile $C_5H_6N_2$ [544-13-8] ZTOMUSMDRMJOTH-UHFFFAOYSA-N | $7.8\times10^{2}$ | 7600 | Plyasunov et al. (2006) | L | |



Table A4.3: Nitriles (C, H, N) (... continued)

| Substance Formula (Trivial Name) [CAS Registry Number] InChIKey | $H_s^{cp}$ (at $T^\ominus$) $\left[\dfrac{\text{mol}}{\text{m}^3\,\text{Pa}}\right]$ | $\dfrac{\text{d}\ln H_s^{cp}}{\text{d}(1/T)}$ [K] | Reference | Type | Note |
|---|---|---|---|---|---|
| hexanedinitrile $C_6H_8N_2$ (adiponitrile) [111-69-3] BTGRAWJCKBQKAO-UHFFFAOYSA-N | $1.5\times10^3$ $8.2\times10^3$ $8.2\times10^3$ $2.4\times10^2$ $2.4\times10^2$ $2.3\times10^1$ $2.2\times10^3$ | 8000 | Plyasunov et al. (2006) Duchowicz et al. (2020) HSDB (2015) Mackay et al. (2006d) Mackay et al. (1995) Duchowicz et al. (2020) Hilal et al. (2008) | L V V V V Q Q | 186 |
| 2-methylpentanedinitrile $C_6H_8N_2$ [4553-62-2] FPPLREPCQJZDAQ-UHFFFAOYSA-N | $3.3\times10^2$ | | HSDB (2015) | Q | 99 |
| tetramethylbutanedinitrile $C_8H_{12}N_2$ [3333-52-6] ZVQXQPNJHRNGID-UHFFFAOYSA-N | $1.9\times10^2$ | | HSDB (2015) | Q | 99 |
| 2-propenenitrile $C_3H_3N$ (acrylonitrile) [107-13-1] NLHHRLWOUZZQLW-UHFFFAOYSA-N | $1.0\times10^{-1}$ $1.2\times10^{-1}$ $7.6\times10^{-2}$ $3.1\times10^{-2}$ $1.3\times10^{-1}$ $8.2\times10^{-2}$ $9.1\times10^{-2}$ $1.3\times10^{-1}$ $9.8\times10^{-2}$ $1.1\times10^{-1}$ $1.1\times10^{-1}$ $1.1\times10^{-1}$ $9.2\times10^{-2}$ $4.3\times10^{-1}$ $2.2\times10^{-2}$ $4.6\times10^{-1}$ $7.2\times10^{-2}$ $9.0\times10^{-2}$ $1.0\times10^{-1}$ | 3900 6800 3400 2800 3600 3600 | Brockbank (2013) Hiatt (2013) Hovorka et al. (2002) Welke et al. (1998) Mackay et al. (2006d) Fogg and Sangster (2003) Lide and Frederikse (1995) Mackay et al. (1995) Hwang et al. (1992) Goldstein (1982) Mackay et al. (1995) Ryan et al. (1988) Keshavarz et al. (2022) Duchowicz et al. (2020) Hilal et al. (2008) Modarresi et al. (2007) Kühne et al. (2005) Duchowicz et al. (2020) Mackay et al. (2006d) Kühne et al. (2005) Yaws (1999) | L M M M V V V V V X C C Q Q Q Q Q ? ? ? ? | 1 38 298 184 67 185, 21 21, 12 |
| 2-butenenitrile $C_4H_5N$ (crotononitrile) [4786-20-3] NKKMVIVFRUYPLQ-NSCUHMNNSA-N | $8.8\times10^{-2}$ $1.5\times10^{-1}$ $2.7\times10^{-1}$ | | Duchowicz et al. (2020) Duchowicz et al. (2020) Modarresi et al. (2007) | V Q Q | 186 67 |





Table A4.3: Nitriles (C, H, N) (... continued)

| Substance Formula (Trivial Name) [CAS Registry Number] InChIKey | $H_s^{cp}$ (at $T^{\ominus}$) $\left[\dfrac{\text{mol}}{\text{m}^3\,\text{Pa}}\right]$ | $\dfrac{\text{d}\ln H_s^{cp}}{\text{d}(1/T)}$ [K] | Reference | Type | Note |
|---|---|---|---|---|---|
| 3-butenenitrile C$_4$H$_5$N (vinylacetonitrile) [109-75-1] SJNALLRHIVGIBI-UHFFFAOYSA-N | $2.0\times10^{-1}$ | 4900 | Brockbank (2013) | L | 1 |
| 2-methyl-2-propene nitrile C$_4$H$_5$N (methacrylonitrile) [126-98-7] GYCMBHHDWRMZGG-UHFFFAOYSA-N | $3.6\times10^{-2}$ | 4200 | Brockbank (2013) | L | 1 |
| | $5.4\times10^{-2}$ | 6700 | Hiatt (2013) | M | |
| | $4.0\times10^{-2}$ | | Duchowicz et al. (2020) | V | 186 |
| | $4.0\times10^{-2}$ | | HSDB (2015) | V | |
| | $1.2\times10^{-1}$ | | Duchowicz et al. (2020) | Q | |
| | $1.7\times10^{-2}$ | | Hilal et al. (2008) | Q | |
| | $1.6\times10^{-1}$ | | Modarresi et al. (2007) | Q | 67 |
| | | 4000 | Kühne et al. (2005) | Q | |
| | | 4600 | Kühne et al. (2005) | ? | |
| benzenenitrile C$_6$H$_5$CN (benzonitrile) [100-47-0] JFDZBHWFFUWGJE-UHFFFAOYSA-N | $3.5\times10^{-1}$ | 5600 | Brockbank (2013) | L | 1 |
| | $2.9\times10^{-1}$ | 5100 | Lee et al. (2013) | M | |
| | $1.9\times10^{-1}$ | | Duchowicz et al. (2020) | V | 186 |
| | $1.9\times10^{-1}$ | | HSDB (2015) | V | |
| | $1.9\times10^{-1}$ | | Mackay et al. (2006d) | V | |
| | $3.9\times10^{-1}$ | | Schüürmann (2000) | V | |
| | $1.9\times10^{-1}$ | | Mackay et al. (1995) | V | |
| | $1.9\times10^{-1}$ | | Mackay et al. (1995) | V | |
| | $5.0\times10^{-1}$ | | Abraham et al. (1994a) | R | |
| | $2.3\times10^{-1}$ | | Duchowicz et al. (2020) | Q | |
| | $1.3\times10^{-1}$ | | Li et al. (2014) | Q | 241 |
| | $2.4\times10^{-1}$ | | Hilal et al. (2008) | Q | |
| | $6.7\times10^{-1}$ | | Modarresi et al. (2007) | Q | 67 |
| | | 5900 | Kühne et al. (2005) | Q | |
| | $5.2\times10^{-1}$ | | Yaffe et al. (2003) | Q | 248, 249 |
| | 2.3 | | Katritzky et al. (1998) | Q | |
| | $1.5\times10^{-1}$ | | Nirmalakhandan et al. (1997) | Q | |
| | | 6400 | Kühne et al. (2005) | ? | |
| | $9.6\times10^{-1}$ | | Yaws (1999) | ? | 21, 415 |
| | $1.8\times10^{-2}$ | | Yaws and Yang (1992) | ? | 21, 415 |
| | $4.1\times10^{-1}$ | | Abraham et al. (1990) | ? | |
| 2-pyridinecarbonitrile C$_6$H$_4$N$_2$ [100-70-9] FFNVQNRYTPFDDP-UHFFFAOYSA-N | $1.4\times10^2$ | | HSDB (2015) | Q | 99 |
| 3,3'-iminobispropanenitrile C$_6$H$_9$N$_3$ [111-94-4] SBAJRGRUGUQKAF-UHFFFAOYSA-N | $2.0\times10^6$ | | HSDB (2015) | Q | 99 |



Table A4.3: Nitriles (C, H, N) (... continued)

| Substance Formula (Trivial Name) [CAS Registry Number] InChIKey | $H_s^{cp}$ (at $T^\ominus$) $\left[\dfrac{\mathrm{mol}}{\mathrm{m^3\,Pa}}\right]$ | $\dfrac{\mathrm{d\ln} H_s^{cp}}{\mathrm{d}(1/T)}$ [K] | Reference | Type | Note |
|---|---|---|---|---|---|
| 2-methylbenzonitrile $C_8H_7N$ ($o$-tolunitrile) [529-19-1] NWPNXBQSRGKSJB-UHFFFAOYSA-N | $7.6 \times 10^{-1}$ | | Schüürmann (2000) | V | |
| 3-methylbenzonitrile $C_8H_7N$ ($m$-tolunitrile) [620-22-4] BOHCMQZJWOGWTA-UHFFFAOYSA-N | $1.7 \times 10^{-1}$ $3.4 \times 10^{-1}$ $8.8 \times 10^{-1}$ $1.8 \times 10^{-1}$ | | Zhang et al. (2010) Zhang et al. (2010) Zhang et al. (2010) Zhang et al. (2010) | Q Q Q Q | 287, 288 287, 289 287, 290 287, 291 |
| benzeneacetonitrile $C_8H_7N$ (phenylacetonitrile) [140-29-4] SUSQOBVLVYHIEX-UHFFFAOYSA-N | 1.1 $7.0 \times 10^{-2}$ $1.0 \times 10^{1}$ | 6800 6200 5100 | Brockbank (2013) HSDB (2015) Hilal et al. (2008) Kühne et al. (2005) Kühne et al. (2005) | L V Q Q ? | 1 |
| 1,2-benzenedicarbonitrile $C_8H_4N_2$ [91-15-6] XQZYPMVTSDWCCE-UHFFFAOYSA-N | $2.0 \times 10^{1}$ | | HSDB (2015) | Q | 99 |
| 3,7-dimethyl-2,6-octadienenitrile $C_{10}H_{15}N$ (geranyl nitrile) [5146-66-7] HLCSDJLATUNSSI-JXMROGBWSA-N | $2.9 \times 10^{-2}$ | | Helburn et al. (2008) | M | |
| 2,2'-azobis(2-methylbutyronitrile) $C_{10}H_{16}N_4$ [13472-08-7] AVTLBBWTUPQRAY-UHFFFAOYSA-N | $4.5 \times 10^{4}$ $9.2 \times 10^{1}$ $1.5 \times 10^{1}$ $4.4 \times 10^{-1}$ | | Zhang et al. (2010) Zhang et al. (2010) Zhang et al. (2010) Zhang et al. (2010) | Q Q Q Q | 287, 288 287, 289 287, 290 287, 291 |





### A4.4 Amines, amides, amino acids (C, H, O, N)

Table A4.4: Amines, amides, amino acids (C, H, O, N)

| Substance Formula (Trivial Name) [CAS Registry Number] InChIKey | $H_s^{cp}$ (at $T^\ominus$) $\left[\dfrac{\text{mol}}{\text{m}^3\,\text{Pa}}\right]$ | $\dfrac{\text{d}\ln H_s^{cp}}{\text{d}(1/T)}$ [K] | Reference | Type | Note |
|---|---|---|---|---|---|
| formamide $CH_3NO$ [75-12-7] ZHNUHDYFZUAESO-UHFFFAOYSA-N | $7.1\times10^3$ $7.0\times10^3$ $2.3\times10^2$ | | Duchowicz et al. (2020) HSDB (2015) Duchowicz et al. (2020) | V V Q | 186 |
| methyl nitrite $CH_3ONO$ [624-91-9] BLLFVUPNHCTMSV-UHFFFAOYSA-N | $1.5\times10^{-1}$ | | HSDB (2015) | Q | 99 |
| urea $CH_4N_2O$ [57-13-6] XSQUKJJJFZCRTK-UHFFFAOYSA-N | $5.7\times10^6$ $5.7\times10^6$ $1.0\times10^7$ $1.0\times10^7$ $1.1\times10^5$ | | Duchowicz et al. (2020) HSDB (2015) Mackay et al. (2006d) Mackay et al. (1995) Duchowicz et al. (2020) | V V V V Q | 186 |
| ethanolamine $HOC_2H_4NH_2$ [141-43-5] HZAXFHJVJLSVMW-UHFFFAOYSA-N | $4.2\times10^3$ $6.0\times10^4$ $1.1\times10^4$ $9.7\times10^2$ | 8300 5800 | Kim et al. (2008) Bone et al. (1983) Modarresi et al. (2007) Nguyen (2013) | M M Q ? | 550 12 67 565, 11 |
| 1,1'-azodiformamide $C_2H_4N_4O_2$ [123-77-3] XOZUGNYVDXMRKW-UHFFFAOYSA-N | $1.2\times10^7$ | | HSDB (2015) | V | |
| ethyl nitrite $C_2H_5ONO$ [109-95-5] QQZWEECEMNQSTG-UHFFFAOYSA-N | $1.1\times10^{-1}$ | | HSDB (2015) | Q | 99 |
| carbamic acid, methyl ester $C_2H_5NO_2$ [598-55-0] GTCAXTIRRLKXRU-UHFFFAOYSA-N | $2.5\times10^2$ | | HSDB (2015) | Q | 99 |
| acetaldoxime $C_2H_5NO$ (acetaldehyde oxime) [107-29-9] FZENGILVLUJGJX-UHFFFAOYSA-N | $1.7$ | | HSDB (2015) | Q | 447 |



Table A4.4: Amines, amides, amino acids (C, H, O, N) (...continued)

| Substance Formula (Trivial Name) [CAS Registry Number] InChIKey | $H_s^{cp}$ (at $T^{\ominus}$) $\left[\dfrac{\mathrm{mol}}{\mathrm{m}^3\,\mathrm{Pa}}\right]$ | $\dfrac{\mathrm{d}\ln H_s^{cp}}{\mathrm{d}(1/T)}$ [K] | Reference | Type | Note |
|---|---|---|---|---|---|
| ethanamide | $5.3\times10^3$ | | Wolfenden (1976) | M | |
| C$_2$H$_5$NO | $2.8\times10^3$ | | Mackay et al. (2006d) | V | |
| (acetamide) | $2.8\times10^3$ | | Mackay et al. (1995) | V | |
| [60-35-5] | $3.8\times10^3$ | | Yaws (2003) | X | 237 |
| DLFVBJFMPXGRIB-UHFFFAOYSA-N | $9.0\times10^2$ | | HSDB (2015) | Q | 99 |
| | $2.8\times10^3$ | | Gharagheizi et al. (2012) | Q | |
| | $3.7\times10^3$ | | Gharagheizi et al. (2010) | Q | 246 |
| | $4.2\times10^3$ | | Hilal et al. (2008) | Q | |
| | $5.1\times10^3$ | | Modarresi et al. (2007) | Q | 67 |
| | $2.2\times10^3$ | | Yaffe et al. (2003) | Q | 248, 249 |
| | $2.2\times10^2$ | | Katritzky et al. (1998) | Q | |
| N-methylmethanamide | $1.4\times10^3$ | 7200 | Burkholder et al. (2019) | L | |
| C$_2$H$_5$NO | $1.4\times10^3$ | 7200 | Burkholder et al. (2015) | L | |
| (N-methylformamide) | $1.4\times10^3$ | 7200 | Brockbank (2013) | L | 1 |
| [123-39-7] | $1.5\times10^3$ | 7600 | Bernauer and Dohnal (2008) | M | 1 |
| ATHHXGZTWNVVOU-UHFFFAOYSA-N | $5.0\times10^2$ | | Duchowicz et al. (2020) | V | 186 |
| | $4.9\times10^2$ | | HSDB (2015) | V | |
| | $9.7\times10^1$ | | Duchowicz et al. (2020) | Q | |
| | $5.6\times10^2$ | | Hilal et al. (2008) | Q | |
| | $1.6\times10^2$ | | Modarresi et al. (2007) | Q | 67 |
| N-nitrosodimethylamine | 3.9 | | Burkholder et al. (2019) | L | |
| C$_2$H$_6$N$_2$O | 3.9 | | Burkholder et al. (2015) | L | |
| [62-75-9] | $2.9\times10^{-1}$ | 13000 | Thompson et al. (2018) | M | |
| UMFJAHHVKNCGLG-UHFFFAOYSA-N | 4.1 | | Haruta et al. (2011) | M | 12 |
| | 6.1 | 6400 | Klein (1982) | M | |
| | 5.2 | | Mirvish et al. (1976) | M | 14 |
| | $3.0\times10^{-1}$ | | Mackay et al. (1995) | C | |
| | 9.5 | | Hilal et al. (2008) | Q | |
| | 3.6 | | Modarresi et al. (2007) | Q | 67 |
| | $3.0\times10^{-1}$ | | Mackay et al. (2006d) | ? | |
| methylnitrosourea | $>1.9\times10^2$ | | Mirvish et al. (1976) | M | 14 |
| C$_2$H$_5$N$_3$O$_2$ | | | | | |
| [684-93-5] | | | | | |
| ZRKWMRDKSOPRRS-UHFFFAOYSA-N | | | | | |
| ethanediamide | $5.0\times10^5$ | | Duchowicz et al. (2020) | V | 186 |
| C$_2$H$_4$N$_2$O$_2$ | $2.2\times10^6$ | | Duchowicz et al. (2020) | Q | |
| [471-46-5] | | | | | |
| YIKSCQDJHCMVMK-UHFFFAOYSA-N | | | | | |
| nitrosoazetidine | $>1.9\times10^2$ | | Mirvish et al. (1976) | M | 14 |
| C$_3$H$_6$N$_2$O | | | | | |
| [15216-10-1] | | | | | |
| SNKTZBNDUVWOAZ-UHFFFAOYSA-N | | | | | |



Table A4.4: Amines, amides, amino acids (C, H, O, N) (...continued)

| Substance Formula (Trivial Name) [CAS Registry Number] InChIKey | $H_s^{cp}$ (at $T^\ominus$) $\left[\dfrac{\text{mol}}{\text{m}^3\,\text{Pa}}\right]$ | $\dfrac{\text{d}\ln H_s^{cp}}{\text{d}(1/T)}$ [K] | Reference | Type | Note |
|---|---|---|---|---|---|
| methylnitrosoacetamide $C_3H_6N_2O_2$ [7417-67-6] FFLFWNRFMZRFKU-UHFFFAOYSA-N | $8.6\times10^{-2}$ | | Mirvish et al. (1976) | M | 14 |
| ethylnitrosocyanamide $C_3H_5N_3O$ [38434-77-4] LMIMSGCBKHFTDY-UHFFFAOYSA-N | $2.6\times10^{-1}$ | | Mirvish et al. (1976) | M | 14 |
| 2-propenamide $C_3H_5NO$ (acrylamide) [79-06-1] HRPVXLWXLXDGHG-UHFFFAOYSA-N | $5.8\times10^{3}$ $5.5\times10^{3}$ $6.9\times10^{3}$ $3.1\times10^{4}$ $6.9\times10^{3}$ $2.9\times10^{4}$ $4.1\times10^{2}$ $7.3\times10^{2}$ $2.9\times10^{4}$ $4.1\times10^{3}$ $7.3\times10^{3}$ | 8400 7900 | Duchowicz et al. (2020) HSDB (2015) Mackay et al. (2006d) Lide and Frederikse (1995) Mackay et al. (1995) Yaws (2003) Duchowicz et al. (2020) Gharagheizi et al. (2012) Gharagheizi et al. (2010) Hilal et al. (2008) Modarresi et al. (2007) Kühne et al. (2005) Kühne et al. (2005) | V V V V V X Q Q Q Q Q Q ? | 186 237, 38 246 67 |
| methylvinylnitrosamine $C_3H_6N_2O$ (N-nitrosomethylvinylamine) [4549-40-0] AWZVYNHQGTZJIH-UHFFFAOYSA-N | 2.7 | | HSDB (2015) | Q | 99 |
| urethane $C_3H_7NO_2$ [51-79-6] JOYRKODLDBILNP-UHFFFAOYSA-N | $1.5\times10^{2}$ $1.5\times10^{2}$ $2.5\times10^{1}$ $1.1\times10^{1}$ $6.4\times10^{1}$ $7.3\times10^{1}$ | | Duchowicz et al. (2020) HSDB (2015) Duchowicz et al. (2020) Hilal et al. (2008) Modarresi et al. (2007) Katritzky et al. (1998) | V V Q Q Q Q | 186 67 |
| propanamide $C_3H_7NO$ (propionamide) [79-05-0] QLNJFJADRCOGBJ-UHFFFAOYSA-N | $3.3\times10^{3}$ | 8800 | Plyasunov et al. (2001) | T | |
| N,N-dimethylmethanamide $C_3H_7NO$ (N,N-dimethylformamide) [68-12-2] ZMXDDKWLCZADIW-UHFFFAOYSA-N | $1.3\times10^{2}$ $1.3\times10^{2}$ $1.4\times10^{2}$ $1.6\times10^{2}$ $2.2\times10^{2}$ $4.5\times10^{1}$ 5.8 | 6600 6600 7700 7500 | Burkholder et al. (2019) Burkholder et al. (2015) Brockbank (2013) Bernauer and Dohnal (2008) Abraham et al. (1994a) Hilal et al. (2008) Modarresi et al. (2007) | L L L M R Q Q | 1, 566 1 67 |





Table A4.4: Amines, amides, amino acids (C, H, O, N) (... continued)

| Substance Formula (Trivial Name) [CAS Registry Number] InChIKey | $H_s^{cp}$ (at $T^\ominus$) $\left[\dfrac{\text{mol}}{\text{m}^3\,\text{Pa}}\right]$ | $\dfrac{\text{d}\ln H_s^{cp}}{\text{d}(1/T)}$ [K] | Reference | Type | Note |
|---|---|---|---|---|---|
| | $2.6\times10^1$ | | Katritzky et al. (1998) | Q | |
| | $2.2\times10^2$ | | Nirmalakhandan et al. (1997) | Q | |
| | $1.6\times10^2$ | | Taft et al. (1985) | Q | |
| N-methylacetamide $C_3H_7NO$ [79-16-3] OHLUUHNLEMFGTQ-UHFFFAOYSA-N | $2.1\times10^3$ $2.1\times10^3$ $3.2\times10^3$ $2.3\times10^2$ $2.3\times10^2$ $3.8\times10^1$ $1.4\times10^2$ | 7600 7600 8900 | Burkholder et al. (2019) Burkholder et al. (2015) Bernauer and Dohnal (2008) Duchowicz et al. (2020) HSDB (2015) Duchowicz et al. (2020) Modarresi et al. (2007) | L L M V V Q Q | 1 186 67 |
| N-methyl-N-nitrosoethanamine $C_3H_8N_2O$ (N-nitrosomethylethylamine) [10595-95-6] RTDCJKARQCRONF-UHFFFAOYSA-N | 6.9 | | HSDB (2015) | Q | 447 |
| 2-methoxyethanamine $C_3H_9NO$ (2-methoxyethylamine) [109-85-3] ASUDFOJKTJLAIK-UHFFFAOYSA-N | $2.5\times10^1$ | 7600 | Cabani et al. (1978) | T | |
| 2-(methylamino)ethanol $C_3H_9NO$ [109-83-1] OPKOKAMJFNKNAS-UHFFFAOYSA-N | $9.0\times10^1$ | | HSDB (2015) | V | |
| 1-amino-2-propanol $C_3H_9NO$ [78-96-6] HXKKHQJGJAFBHI-UHFFFAOYSA-N | $4.2\times10^4$ | | HSDB (2015) | Q | 545 |
| 2-amino-1-propanol $C_3H_9NO$ (alaninol) [6168-72-5] BKMMTJMQCTUHRP-UHFFFAOYSA-N | $1.3\times10^3$ | 7700 | Nguyen (2013) | M | 11 |
| N-nitroso-N-methylurethane $C_4H_8N_2O_3$ (N-nitroso-N-methylurethane) [615-53-2] CAUBWLYZCDDYEF-UHFFFAOYSA-N | $3.9\times10^{-1}$ 1.8 1.8 $1.6\times10^1$ $3.2\times10^1$ | | Mirvish et al. (1976) Duchowicz et al. (2020) HSDB (2015) Duchowicz et al. (2020) Modarresi et al. (2007) | M V V Q Q | 14 186 67 |
| dinitrosopiperazine $C_4H_8N_4O_2$ [140-79-4] WNSYEWGYAFFSSQ-UHFFFAOYSA-N | $>1.9\times10^2$ | | Mirvish et al. (1976) | M | 14 |





790            Rolf Sander: Compilation of Henry's law constants

Table A4.4: Amines, amides, amino acids (C, H, O, N) (... continued)

| Substance Formula (Trivial Name) [CAS Registry Number] InChIKey | $H_s^{cp}$ (at $T^{\ominus}$) $\left[\dfrac{\mathrm{mol}}{\mathrm{m^3\,Pa}}\right]$ | $\dfrac{\mathrm{d}\ln H_s^{cp}}{\mathrm{d}(1/T)}$ [K] | Reference | Type | Note |
|---|---|---|---|---|---|
| 2-amino-3(methylamino)propionic acid $C_4H_{10}N_2O_2$ (3-(methylamino)-(DL)-alanine) [16676-91-8] UJVHVMNGOZXSOZ-UHFFFAOYSA-N | $2.9\times10^7$ | | HSDB (2015) | Q | 99 |
| N-nitrosodiethanolamine $C_4H_{10}N_2O_3$ [1116-54-7] YFCDLVPYFMHRQZ-UHFFFAOYSA-N | $2.0\times10^6$ | | HSDB (2015) | Q | 447 |
| N-nitrosodiethylamine $C_4H_{10}N_2O$ [55-18-5] WBNQDOYYEUMPFS-UHFFFAOYSA-N | $1.2\times10^{-2}$ 5.6 1.4 $7.1\times10^{-1}$ 7.5 3.9 2.6 2.7 | 3800 6300 | Thompson et al. (2018) Klein (1982) Mirvish et al. (1976) Keshavarz et al. (2022) Duchowicz et al. (2020) Hilal et al. (2008) Modarresi et al. (2007) Duchowicz et al. (2020) | M M M Q Q Q Q ? | 14 184 67 185, 21 |
| diethanolamine $C_4H_{11}NO_2$ [111-42-2] ZBCBWPMODOFKDW-UHFFFAOYSA-N | $3.3\times10^3$ $2.6\times10^5$ $2.5\times10^5$ $1.0\times10^4$ | 1300 | Nguyen (2013) Duchowicz et al. (2020) HSDB (2015) Duchowicz et al. (2020) | M V V Q | 11 186 |
| diglycolamine $C_4H_{11}NO_2$ [929-06-6] GIAFURWZWWWBQT-UHFFFAOYSA-N | $6.1\times10^3$ | 3700 | Nguyen (2013) | M | 33, 11 |
| 3-methoxy-1-propanamine $C_4H_{11}NO$ (3-methoxypropylamine) [5332-73-0] FAXDZWQIWUSWJH-UHFFFAOYSA-N | $2.1\times10^1$ $4.8\times10^1$ $4.9\times10^1$ $1.4\times10^1$ | 8700 | Du et al. (2017) Cabani et al. (1978) Du et al. (2017) Du et al. (2017) | M T Q Q | 478 549 |
| 2-amino-2-methyl-1-propanol $C_4H_{11}NO$ [124-68-5] CBTVGIZVANVGBH-UHFFFAOYSA-N | $2.1\times10^2$ $7.0\times10^2$ $1.8\times10^2$ $9.3\times10^1$ | 8500 | Du et al. (2017) Nguyen (2013) Du et al. (2017) Du et al. (2017) | M M Q Q | 478 11 549 |
| N,N-dimethylaminoethanol $C_4H_{11}NO$ (dimethylethanolamine) [108-01-0] UEEJHVSXFDXPFK-UHFFFAOYSA-N | $9.3\times10^1$ $2.6\times10^1$ $4.4\times10^2$ | 7900 | Nguyen (2013) Duchowicz et al. (2020) Duchowicz et al. (2020) | M V Q | 11 186 |





Table A4.4: Amines, amides, amino acids (C, H, O, N) (...continued)

| Substance Formula (Trivial Name) [CAS Registry Number] InChIKey | $H_s^{cp}$ (at $T^{\ominus}$) $\left[\dfrac{\text{mol}}{\text{m}^3\,\text{Pa}}\right]$ | $\dfrac{\text{d}\ln H_s^{cp}}{\text{d}(1/T)}$ [K] | Reference | Type | Note |
|---|---|---|---|---|---|
| 2-[(2-aminoethyl)amino]ethanol $C_4H_{12}N_2O$ [111-41-1] LHIJANUOQQMGNT-UHFFFAOYSA-N | $9.0\times10^7$ | | HSDB (2015) | Q | 99 |
| tetramethylammonium hydroxide $C_4H_{13}NO$ [75-59-2] WGTYBPLFGIVFAS-UHFFFAOYSA-M | $2.3\times10^{10}$ | | HSDB (2015) | Q | 99 |
| methacrylamide $C_4H_7NO$ [79-39-0] FQPSGWSUVKBHSU-UHFFFAOYSA-N | $8.8\times10^3$ | | Ebert et al. (2023) | ? | 316 |
| acetone cyanohydrin $C_4H_7NO$ [75-86-5] MWFMGBPGAXYFAR-UHFFFAOYSA-N | $8.0\times10^1$ | | HSDB (2015) | V | |
| carbamic acid, 1-methylethyl ester $C_4H_9NO_2$ [1746-77-6] OVPLZYJGTGDFNB-UHFFFAOYSA-N | $1.4\times10^2$ | | HSDB (2015) | Q | 99 |
| propylcarbamate $C_4H_9NO_2$ [627-12-3] YNTOKMNHRPSGFU-UHFFFAOYSA-N | $1.0\times10^2$ | | HSDB (2015) | V | |
| butanamide $C_4H_9NO$ [541-35-5] DNSISZSEWVHGLH-UHFFFAOYSA-N | $3.6\times10^3$ | | Ebert et al. (2023) | ? | 316 |
| N,N-dimethylacetamide $C_4H_9NO$ [127-19-5] FXHOOIRPVKKKFG-UHFFFAOYSA-N | $6.1\times10^1$ $6.1\times10^1$ $4.4\times10^2$ $4.4\times10^2$ $1.7\times10^2$ $1.3\times10^1$ $3.6\times10^2$ | 7800 7800 8000 8600 | Burkholder et al. (2019) Burkholder et al. (2015) Brockbank (2013) Bernauer and Dohnal (2008) Hilal et al. (2008) Modarresi et al. (2007) Taft et al. (1985) | L L L M Q Q Q | 1 1 67 |
| 2-butanone, oxime $C_4H_9NO$ [96-29-7] WHIVNJATOVLWBW-UHFFFAOYSA-N | 8.1 | | HSDB (2015) | V | |





Table A4.4: Amines, amides, amino acids (C, H, O, N) (...continued)

| Substance Formula (Trivial Name) [CAS Registry Number] InChIKey | $H_s^{cp}$ (at $T^{\ominus}$) $\left[\dfrac{\mathrm{mol}}{\mathrm{m^3\,Pa}}\right]$ | $\dfrac{\mathrm{d}\ln H_s^{cp}}{\mathrm{d}(1/T)}$ [K] | Reference | Type | Note |
|---|---|---|---|---|---|
| nitrosoethylurethane $C_5H_{10}N_2O_3$ (N-ethyl-N-nitrosourethane) [614-95-9] RAUQLNDTFONODT-UHFFFAOYSA-N | $5.2\times10^2$ | | HSDB (2015) | Q | 99 |
| N,N-diethylmethanamide $C_5H_{11}NO$ (N,N-diethylformamide) [617-84-5] SUAKHGWARZSWIH-UHFFFAOYSA-N | $1.3\times10^2$ | | Yaffe et al. (2003) | Q | 248, 249 |
| N-nitroso-N-butylurea $C_5H_{11}N_3O_2$ [869-01-2] LSWOCDLIYSKTRX-UHFFFAOYSA-N | $4.3\times10^4$ | | HSDB (2015) | Q | 99 |
| N-methyldiethanolamine $C_5H_{13}NO_2$ [105-59-9] CRVGTESFCCXCTH-UHFFFAOYSA-N | $3.3\times10^3$ $3.9\times10^3$ $4.0\times10^5$ $3.1\times10^5$ $3.2\times10^5$ $1.3\times10^5$ $2.4\times10^3$ $1.9\times10^3$ | 3800 12000 | Du et al. (2017) Nguyen (2013) Kim et al. (2008) Duchowicz et al. (2020) HSDB (2015) Duchowicz et al. (2020) Du et al. (2017) Du et al. (2017) | M M M V V Q Q Q | 478 11 550 186 549 |
| methylbutylnitrosamine $C_5H_{12}N_2O$ [7068-83-9] PKTSCJXWLVREKX-UHFFFAOYSA-N | 1.7 | | Mirvish et al. (1976) | M | 14 |
| 2-(isopropylamino)ethanol $C_5H_{13}NO$ [109-56-8] RILLZYSZSDGYGV-UHFFFAOYSA-N | $1.1\times10^2$ $1.4\times10^1$ $5.4\times10^1$ | | Du et al. (2017) Du et al. (2017) Du et al. (2017) | M Q Q | 478 549 |
| 3-(dimethylamino)-1,2-propanediol $C_5H_{13}NO_2$ [623-57-4] QCMHUGYTOGXZIW-UHFFFAOYSA-N | $6.7\times10^2$ $6.1\times10^3$ $6.4\times10^2$ | | Du et al. (2017) Du et al. (2017) Du et al. (2017) | M Q Q | 478 549 |
| 2-amino-2-ethyl-1,3-propanediol $C_5H_{13}NO_2$ [115-70-8] IOAOAKDONABGPZ-UHFFFAOYSA-N | $7.1\times10^2$ $4.0\times10^4$ $2.7\times10^3$ | | Du et al. (2017) Du et al. (2017) Du et al. (2017) | M Q Q | 478 549 |
| daminozide $C_6H_{11}NO_3$ [1596-84-5] NOQGZXFMHARMLW-UHFFFAOYSA-N | $2.3\times10^4$ $4.1\times10^3$ $7.0\times10^1$ | | Duchowicz et al. (2020) Duchowicz et al. (2020) Maniere et al. (2011) | V Q ? | 186 165 |





Table A4.4: Amines, amides, amino acids (C, H, O, N) (...continued)

| Substance<br>Formula<br>(Trivial Name)<br>[CAS Registry Number]<br>InChIKey | $H_s^{cp}$<br>(at $T^\ominus$)<br>$\left[\dfrac{\text{mol}}{\text{m}^3\,\text{Pa}}\right]$ | $\dfrac{\text{d}\ln H_s^{cp}}{\text{d}(1/T)}$<br><br>[K] | Reference | Type | Note |
|---|---|---|---|---|---|
| methylpentylnitrosamine<br>$C_6H_{14}N_2O$<br>[13256-07-0]<br>KSFCDINBDBFFSI-UHFFFAOYSA-N | 2.0 | | Mirvish et al. (1976) | M | 14 |
| ethylbutylnitrosamine<br>$C_6H_{14}N_2O$<br>[4549-44-4]<br>ZGMCNGHHUQZNIH-UHFFFAOYSA-N | $9.9\times10^{-1}$ | | Mirvish et al. (1976) | M | 14 |
| nitrosohexamethyleneimine<br>$C_6H_{12}N_2O$<br>[932-83-2]<br>UZMVSVHUTOAPTD-UHFFFAOYSA-N | $4.3\times10^{1}$ | | Mirvish et al. (1976) | M | 14 |
| 2,6-dimethylnitrosomorpholine<br>$C_6H_{12}N_2O_2$<br>[1456-28-6]<br>DPYMAXOKJUBANR-UHFFFAOYSA-N | $3.5\times10^{1}$ | | Mirvish et al. (1976) | M | 14 |
| 2,6-dimethyldinitrosopiperazine<br>$C_6H_{12}N_4O_2$<br>[55380-34-2]<br>JIWAGFGPBKDFQN-UHFFFAOYSA-N | $>1.9\times10^{2}$ | | Mirvish et al. (1976) | M | 14 |
| N-(1-methylethyl)-2-propenamide<br>$C_6H_{11}NO$<br>(N-isopropylacrylamide)<br>[2210-25-5]<br>QNILTEGFHQSKFF-UHFFFAOYSA-N | $4.3\times10^{2}$ | | HSDB (2015) | Q | 99 |
| hexanamide<br>$C_6H_{13}NO$<br>[628-02-4]<br>ALBYIUDWACNRRB-UHFFFAOYSA-N | $1.8\times10^{3}$ | | Ebert et al. (2023) | ? | 316 |
| N-butylacetamide<br>$C_6H_{13}NO$<br>[1119-49-9]<br>GYLDXXLJMRTVSS-UHFFFAOYSA-N | $2.7\times10^{3}$<br>$5.2\times10^{2}$ | | Gibbs et al. (1991)<br>Hilal et al. (2008) | M<br>Q | |
| N-(1-methylethyl)-N-nitroso-2-propanamine<br>$C_6H_{14}N_2O$<br>(nitrosodiisopropylamine)<br>[601-77-4]<br>AUIKJTGFPFLMFP-UHFFFAOYSA-N | 1.2<br>$3.4\times10^{-1}$<br>1.1 | | Mirvish et al. (1976)<br>Hilal et al. (2008)<br>Modarresi et al. (2007) | M<br>Q<br>Q | 14<br><br>67 |



Table A4.4: Amines, amides, amino acids (C, H, O, N) (…continued)

| Substance Formula (Trivial Name) [CAS Registry Number] InChIKey | $H_s^{cp}$ (at $T^\ominus$) $\left[\dfrac{\mathrm{mol}}{\mathrm{m^3\,Pa}}\right]$ | $\dfrac{\mathrm{d}\ln H_s^{cp}}{\mathrm{d}(1/T)}$ [K] | Reference | Type | Note |
|---|---|---|---|---|---|
| N-nitrosodipropylamine C$_6$H$_{14}$N$_2$O (N,N-dipropylnitrosamine) [621-64-7] YLKFDHTUAUWZPQ-UHFFFAOYSA-N | 1.8 2.8 2.8 1.6 5.8 1.3 | | Mirvish et al. (1976) Mackay et al. (2006d) Mackay et al. (1995) Mackay et al. (1995) Hilal et al. (2008) Modarresi et al. (2007) | M V V C Q Q | 14 67 |
| N-(2-hydroxyethyl)piperazine C$_6$H$_{14}$N$_2$O [103-76-4] WFCSWCVEJLETKA-UHFFFAOYSA-N | $2.9\times10^3$ | 6400 | Nguyen (2013) | M | 33, 11 |
| N-ethyldiethanolamine C$_6$H$_{15}$NO$_2$ [139-87-7] AKNUHUCEWALCOI-UHFFFAOYSA-N | $9.0\times10^4$ | | HSDB (2015) | Q | 99 |
| triethanolamine C$_6$H$_{15}$NO$_3$ [102-71-6] GSEJCLTVZPLZKY-UHFFFAOYSA-N | $1.4\times10^7$ $1.4\times10^7$ $3.3\times10^7$ | | Duchowicz et al. (2020) HSDB (2015) Duchowicz et al. (2020) | V V Q | 186 |
| $o$-aminophenol C$_6$H$_7$NO (2-aminophenol) [95-55-6] CDAWCLOXVUBKRW-UHFFFAOYSA-N | $4.9\times10^4$ | | HSDB (2015) | Q | 99 |
| 4-aminophenol C$_6$H$_7$NO [123-30-8] PLIKAWJENQZMHA-UHFFFAOYSA-N | $2.7\times10^4$ $2.7\times10^4$ $1.8\times10^4$ | | Duchowicz et al. (2020) HSDB (2015) Duchowicz et al. (2020) | V V Q | 186 |
| 3-aminophenol C$_6$H$_7$NO [591-27-5] CWLKGDAVCFYWJK-UHFFFAOYSA-N | $3.7\times10^4$ | | HSDB (2015) | Q | 99 |
| 4-(2-hydroxyethyl)morpholine C$_6$H$_{13}$NO$_2$ [622-40-2] KKFDCBRMNNSAAW-UHFFFAOYSA-N | $1.4\times10^3$ $6.0\times10^1$ $1.7\times10^3$ | | Du et al. (2017) Du et al. (2017) Du et al. (2017) | M Q Q | 478 549 |
| 4-hydroxy-N-methylpiperidine C$_6$H$_{13}$NO [106-52-5] BAUWRHPMUVYFOD-UHFFFAOYSA-N | $1.3\times10^3$ $4.6\times10^3$ $1.1\times10^3$ | | Du et al. (2017) Du et al. (2017) Du et al. (2017) | M Q Q | 478 549 |



Table A4.4: Amines, amides, amino acids (C, H, O, N) (. . . continued)

| Substance<br>Formula<br>(Trivial Name)<br>[CAS Registry Number]<br>InChIKey | $H_s^{cp}$<br>(at $T^{\ominus}$)<br>$\left[\dfrac{\mathrm{mol}}{\mathrm{m^3\,Pa}}\right]$ | $\dfrac{\mathrm{d}\ln H_s^{cp}}{\mathrm{d}(1/T)}$<br><br>[K] | Reference | Type | Note |
|---|---|---|---|---|---|
| dimethylaminoethoxyethanol<br>$C_6H_{15}NO_2$<br>[1704-62-7]<br>YSAANLSYLSUVHB-UHFFFAOYSA-N | $8.3\times10^2$<br>$4.3\times10^3$<br>$5.6\times10^2$<br>$6.2\times10^2$ | <br>8500 | Du et al. (2017)<br>Nguyen (2013)<br>Du et al. (2017)<br>Du et al. (2017) | M<br>M<br>Q<br>Q | 478<br>11<br>549 |
| 2-(diethylamino)ethanol<br>$C_6H_{15}NO$<br>[100-37-8]<br>BFSVOASYOCHEOV-UHFFFAOYSA-N | 5.9<br>3.2<br>$1.8\times10^1$ | | Du et al. (2017)<br>Du et al. (2017)<br>Du et al. (2017) | M<br>Q<br>Q | 478<br>549 |
| bis(2-methoxyethyl)amine<br>$C_6H_{15}NO_2$<br>[111-95-5]<br>IBZKBSXREAQDTO-UHFFFAOYSA-N | $1.1\times10^2$<br>$2.3\times10^2$<br>$1.5\times10^2$ | | Du et al. (2017)<br>Du et al. (2017)<br>Du et al. (2017) | M<br>Q<br>Q | 478<br>549 |
| cyclohexanone oxime<br>$C_6H_{11}NO$<br>[100-64-1]<br>VEZUQRBDRNJBJY-UHFFFAOYSA-N | $4.3\times10^1$<br>$1.4\times10^1$ | | Duchowicz et al. (2020)<br>Duchowicz et al. (2020) | V<br>Q | 186 |
| $p$-diaminoanisole<br>$C_7H_{10}N_2O$<br>(2-methoxy-1,4-benzenediamine)<br>[5307-02-8]<br>HGUYBLVGLMAUFF-UHFFFAOYSA-N | $2.5\times10^5$ | | HSDB (2015) | Q | 99 |
| 4-methoxy-1,3-benzenediamine<br>$C_7H_{10}N_2O$<br>[615-05-4]<br>BAHPQISAXRFLCL-UHFFFAOYSA-N | $1.4\times10^4$ | | HSDB (2015) | Q | 99 |
| 2-cyano-N-<br>[(ethylamino)carbonyl]-2-<br>(methoxyimino)acetamide<br>$C_7H_{10}N_4O_3$<br>(cymoxanil)<br>[57966-95-7]<br>XERJKGMBORTKEO-UHFFFAOYSA-N | $3.0\times10^4$<br><br><br>$3.0\times10^4$<br>$6.2\times10^2$<br>$9.9\times10^8$<br>$3.2\times10^1$<br>$2.6\times10^4$<br><br>$3.0\times10^4$ | | Duchowicz et al. (2020)<br><br><br>HSDB (2015)<br>Barcelo and Hennion (1997)<br>Duchowicz et al. (2020)<br>Goodarzi et al. (2010)<br>Maniere et al. (2011)<br><br>Maniere et al. (2011) | V<br><br><br>V<br>X<br>Q<br>Q<br>?<br><br>? | 186<br><br><br><br>567<br><br>568, 569<br>241, 493,<br>165<br>241, 570,<br>165 |
| isocyanatocyclohexane<br>$C_7H_{11}NO$<br>[3173-53-3]<br>KQWGXHWJMSMDJJ-UHFFFAOYSA-N | $5.8\times10^{-3}$ | | HSDB (2015) | Q | 99 |



Table A4.4: Amines, amides, amino acids (C, H, O, N) (... continued)

| Substance Formula (Trivial Name) [CAS Registry Number] InChIKey | $H_s^{cp}$ (at $T^\ominus$) $\left[\dfrac{\text{mol}}{\text{m}^3\,\text{Pa}}\right]$ | $\dfrac{\text{d}\ln H_s^{cp}}{\text{d}(1/T)}$ [K] | Reference | Type | Note |
|---|---|---|---|---|---|
| $L$-theanine $C_7H_{14}N_2O_3$ [3081-61-6] DATAGRPVKZEWHA-YFKPBYRVSA-N | $1.1\times10^{10}$ | | HSDB (2015) | Q | 447 |
| tetryl $C_7H_5N_5O_8$ [479-45-8] AGUIVNYEYSCPNI-UHFFFAOYSA-N | $3.7\times10^{3}$ | | HSDB (2015) | Q | 99 |
| anthranilic acid $C_7H_7NO_2$ [118-92-3] RWZYAGGXGHYGMB-UHFFFAOYSA-N | $2.6\times10^{5}$ | | HSDB (2015) | Q | 99 |
| 3-aminobenzoic acid $C_7H_7NO_2$ [99-05-8] XFDUHJPVQKIXHO-UHFFFAOYSA-N | $3.7\times10^{6}$ | | Ebert et al. (2023) | ? | 316 |
| 4-aminobenzoic acid $C_7H_7NO_2$ [150-13-0] ALYNCZNDIQEVRV-UHFFFAOYSA-N | $6.6\times10^{4}$ | | HSDB (2015) | V | |
| salicylamide $C_7H_7NO_2$ [65-45-2] SKZKKFZAGNVIMN-UHFFFAOYSA-N | $2.0\times10^{4}$ $3.4\times10^{4}$ | | Abraham et al. (2019) HSDB (2015) | Q Q | 99 |
| mesalamine $C_7H_7NO_3$ [89-57-6] KBOPZPXVLCULAV-UHFFFAOYSA-N | $2.0\times10^{6}$ | | HSDB (2015) | Q | 99 |
| N-phenylformamide $C_7H_7NO$ [103-70-8] DYDNPESBYVVLBO-UHFFFAOYSA-N | $1.2\times10^{3}$ | | HSDB (2015) | Q | 99 |
| benzamide $C_7H_7NO$ [55-21-0] KXDAEFPNCMNJSK-UHFFFAOYSA-N | $2.2\times10^{4}$ $4.7\times10^{4}$ $5.2\times10^{3}$ $9.0\times10^{2}$ $3.8\times10^{3}$ $2.7\times10^{4}$ $8.2\times10^{3}$ $4.0\times10^{4}$ $4.0\times10^{4}$ | | Mackay et al. (2006d) Mackay et al. (1995) Abraham et al. (1994a) Keshavarz et al. (2022) Duchowicz et al. (2020) Hilal et al. (2008) Modarresi et al. (2007) Nirmalakhandan et al. (1997) Duchowicz et al. (2020) HSDB (2015) | V V R Q Q Q Q Q ? ? | 558 299 67 185, 21 419 |



Table A4.4: Amines, amides, amino acids (C, H, O, N) (...continued)

| Substance Formula (Trivial Name) [CAS Registry Number] InChIKey | $H_s^{cp}$ (at $T^{\ominus}$) $\left[\dfrac{\mathrm{mol}}{\mathrm{m^3\,Pa}}\right]$ | $\dfrac{\mathrm{d}\ln H_s^{cp}}{\mathrm{d}(1/T)}$ [K] | Reference | Type | Note |
|---|---|---|---|---|---|
| anthranilamide $C_7H_8N_2O$ (2-aminobenzamide) [88-68-6] PXBFMLJZNCDSMP-UHFFFAOYSA-N | $1.3 \times 10^7$ | | HSDB (2015) | Q | 99 |
| N-methyl-N-nitrosobenzenamine $C_7H_8N_2O$ [614-00-6] MAXCWSIJKVASQC-UHFFFAOYSA-N | 2.0 | | HSDB (2015) | Q | 99 |
| 2-methoxy-benzenamine $C_7H_9NO$ (2-methoxyaniline) [90-04-0] VMPITZXILSNTON-UHFFFAOYSA-N | $1.1 \times 10^1$ $1.2 \times 10^1$ $6.7 \times 10^1$ $1.1 \times 10^1$ $2.8 \times 10^1$ 5.6 $1.5 \times 10^1$ | | Duchowicz et al. (2020) Abraham et al. (1994a) Duchowicz et al. (2020) HSDB (2015) Hilal et al. (2008) Modarresi et al. (2007) Nirmalakhandan et al. (1997) | V R Q Q Q Q Q | 186 99 67 |
| 3-methoxy-benzenamine $C_7H_9NO$ (3-methoxyaniline) [536-90-3] NCBZRJODKRCREW-UHFFFAOYSA-N | $9.0 \times 10^1$ $9.0 \times 10^1$ $1.8 \times 10^2$ $2.8 \times 10^1$ $6.4 \times 10^1$ $1.5 \times 10^1$ | | Abraham et al. (1994a) HSDB (2015) Hilal et al. (2008) Modarresi et al. (2007) English and Carroll (2001) Nirmalakhandan et al. (1997) | R Q Q Q Q Q | 99 67 230, 231 |
| 4-methoxy-benzenamine $C_7H_9NO$ (4-methoxyaniline) [104-94-9] BHAAPTBBJKJZER-UHFFFAOYSA-N | $1.5 \times 10^2$ $1.2 \times 10^2$ $1.8 \times 10^2$ $5.8 \times 10^1$ $1.4 \times 10^2$ $2.8 \times 10^1$ $9.9 \times 10^1$ $1.5 \times 10^1$ $1.5 \times 10^2$ | | Altschuh et al. (1999) Abraham et al. (1994a) Keshavarz et al. (2022) Duchowicz et al. (2020) Hilal et al. (2008) Modarresi et al. (2007) English and Carroll (2001) Nirmalakhandan et al. (1997) Duchowicz et al. (2020) | M R Q Q Q Q Q Q ? | 67 230, 231 185, 21 |
| 4-(methylamino)phenol $C_7H_9NO$ (N-methyl-4-aminophenol) [150-75-4] ZFIQGRISGKSVAG-UHFFFAOYSA-N | $2.2 \times 10^4$ | | HSDB (2015) | Q | 99 |
| 3-(diethylamino)-1,2-propanediol $C_7H_{17}NO_2$ [621-56-7] LTACQVCHVAUOKN-UHFFFAOYSA-N | $5.4 \times 10^2$ $7.0 \times 10^2$ $2.6 \times 10^2$ | | Du et al. (2017) Du et al. (2017) Du et al. (2017) | M Q Q | 478 549 |



Table A4.4: Amines, amides, amino acids (C, H, O, N) (. . . continued)

| Substance<br>Formula<br>(Trivial Name)<br>[CAS Registry Number]<br>InChIKey | $H_s^{cp}$<br>(at $T^{\ominus}$)<br>$\left[\dfrac{\text{mol}}{\text{m}^3\,\text{Pa}}\right]$ | $\dfrac{\text{d}\ln H_s^{cp}}{\text{d}(1/T)}$<br><br>[K] | Reference | Type | Note |
|---|---|---|---|---|---|
| 1-(2-hydroxyethyl)piperidine<br>$C_7H_{15}NO$<br>(2-piperidinoethanol)<br>[3040-44-6]<br>KZTWONRVIPPDKH-UHFFFAOYSA-N | $5.1\times10^1$<br>$6.1\times10^{-1}$<br>$4.9\times10^1$ | | Du et al. (2017)<br>Du et al. (2017)<br>Du et al. (2017) | M<br>Q<br>Q | 478<br>549 |
| 2-piperidineethanol<br>$C_7H_{15}NO$<br>[1484-84-0]<br>PTHDBHDZSMGHKF-UHFFFAOYSA-N | $3.9\times10^2$<br>$1.2\times10^2$<br>$2.3\times10^2$ | | Du et al. (2017)<br>Du et al. (2017)<br>Du et al. (2017) | M<br>Q<br>Q | 478<br>549 |
| emylcamate<br>$C_7H_{15}NO_2$<br>[78-28-4]<br>SLWGJZPKHAXZQL-UHFFFAOYSA-N | $3.0\times10^{-1}$<br>8.7 | | Duchowicz et al. (2020)<br>Duchowicz et al. (2020) | V<br>Q | 186 |
| defenuron<br>$C_8H_{10}N_2O$<br>[1007-36-9]<br>SQBHGDSDVWCPHN-UHFFFAOYSA-N | $8.5\times10^5$ | | MacBean (2012a) | ? | 12 |
| N-methyl-N-nitrosobenzenemethanamine<br>$C_8H_{10}N_2O$<br>[937-40-6]<br>NGXUJKBJBFLCAR-UHFFFAOYSA-N | $7.9\times10^{-1}$<br><br>$1.2\times10^2$<br>$3.8\times10^1$ | | Mirvish et al. (1976)<br><br>Hilal et al. (2008)<br>Modarresi et al. (2007) | M<br><br>Q<br>Q | 14<br><br><br>67 |
| norepinephrine<br>$C_8H_{11}NO_3$<br>[51-41-2]<br>SFLSHLFXELFNJZ-QMMMGPOBSA-N | $3.1\times10^{13}$ | | HSDB (2015) | Q | 99 |
| 2,4-dimethoxyaniline<br>$C_8H_{11}NO_2$<br>[2735-04-8]<br>GEQNZVKIDIPGCO-UHFFFAOYSA-N | $1.6\times10^1$ | | Ebert et al. (2023) | ? | 318 |
| 4-methoxy-2-methylbenzenamine<br>$C_8H_{11}NO$<br>($m$-cresidine)<br>[102-50-1]<br>CDGNLUSBENXDGG-UHFFFAOYSA-N | $8.2\times10^1$ | | HSDB (2015) | Q | 99 |
| $p$-cresidine<br>$C_8H_{11}NO$<br>[120-71-8]<br>WXWCDTXEKCVRRO-UHFFFAOYSA-N | $8.0\times10^1$ | | HSDB (2015) | Q | 99 |



Table A4.4: Amines, amides, amino acids (C, H, O, N) (... continued)

| Substance<br>Formula<br>(Trivial Name)<br>[CAS Registry Number]<br>InChIKey | $H_s^{cp}$<br>(at $T^\ominus$)<br>$\left[\dfrac{\mathrm{mol}}{\mathrm{m^3\,Pa}}\right]$ | $\dfrac{\mathrm{d}\ln H_s^{cp}}{\mathrm{d}(1/T)}$<br><br>[K] | Reference | Type | Note |
|---|---|---|---|---|---|
| N-nitrosodi-N-butylamine<br>$C_8H_{18}N_2O$<br>[924-16-3]<br>YGJHZCLPZAZIHH-UHFFFAOYSA-N | $7.2\times10^{-1}$<br>$7.5\times10^{-1}$<br>$1.1$ | | Mirvish et al. (1976)<br>Hilal et al. (2008)<br>Modarresi et al. (2007) | M<br>Q<br>Q | 14<br><br>67 |
| (diisopropylamino)-ethanol<br>$C_8H_{19}NO$<br>[96-80-0]<br>ZYWUVGFIXPNBDL-UHFFFAOYSA-N | $2.2$<br>$2.3$<br>$2.4$<br>$1.9\times10^2$<br>$8.2\times10^1$ | | Du et al. (2017)<br>Du et al. (2017)<br>Du et al. (2017)<br>Hilal et al. (2008)<br>Modarresi et al. (2007) | M<br>Q<br>Q<br>Q<br>Q | 478<br>549<br><br><br>67 |
| phthalamide<br>$C_8H_8N_2O_2$<br>[88-96-0]<br>NAYYNDKKHOIIOD-UHFFFAOYSA-N | $7.0\times10^6$ | | HSDB (2015) | Q | 99 |
| acetaminophen<br>$C_8H_9NO_2$<br>[103-90-2]<br>RZVAJINKPMORJF-UHFFFAOYSA-N | $1.5\times10^7$ | | HSDB (2015) | Q | 99 |
| methyl anthranilate<br>$C_8H_9NO_2$<br>[134-20-3]<br>VAMXMNNIEUEQDV-UHFFFAOYSA-N | $5.2$<br>$5.2$<br>$3.6$<br>$2.6\times10^2$ | | Duchowicz et al. (2020)<br>HSDB (2015)<br>Dupeux et al. (2022)<br>Duchowicz et al. (2020) | V<br>V<br>Q<br>Q | 186<br><br>259 |
| 4'-aminoacetophenone<br>$C_8H_9NO$<br>(4-acetylaniline)<br>[99-92-3]<br>GPRYKVSEZCQIHD-UHFFFAOYSA-N | $2.2\times10^3$ | | HSDB (2015) | Q | 99 |
| N-phenylacetamide<br>$C_8H_9NO$<br>(acetanilide)<br>[103-84-4]<br>FZERHIULMFGESH-UHFFFAOYSA-N | $4.7\times10^2$<br>$1.5\times10^3$<br>$1.6\times10^3$ | | Yaws (2003)<br>Dupeux et al. (2022)<br>HSDB (2015) | X<br>Q<br>Q | 258<br>259<br>99 |
| tropine<br>$C_8H_{15}NO$<br>[120-29-6]<br>CYHOMWAPJJPNMW-UHFFFAOYSA-N | $1.3\times10^3$<br>$2.6\times10^3$<br>$5.6\times10^2$ | | Du et al. (2017)<br>Du et al. (2017)<br>Du et al. (2017) | M<br>Q<br>Q | 478<br>549 |
| bis[2-(N,N-dimethylamino)ethyl]<br>ether<br>$C_8H_{20}N_2O$<br>[3033-62-3]<br>GTEXIOINCJRBIO-UHFFFAOYSA-N | $7.9\times10^1$<br><br>$2.4\times10^2$<br>$7.9\times10^1$ | | Du et al. (2017)<br><br>Du et al. (2017)<br>Du et al. (2017) | M<br><br>Q<br>Q | 478<br><br>549 |




Table A4.4: Amines, amides, amino acids (C, H, O, N) (... continued)

| Substance Formula (Trivial Name) [CAS Registry Number] InChIKey | $H_s^{cp}$ (at $T^{\ominus}$) $\left[\dfrac{\mathrm{mol}}{\mathrm{m}^3\,\mathrm{Pa}}\right]$ | $\dfrac{\mathrm{d}\ln H_s^{cp}}{\mathrm{d}(1/T)}$ [K] | Reference | Type | Note |
|---|---|---|---|---|---|
| methylcarbamic acid, 3-methylphenyl ester $C_9H_{11}NO_2$ (metolcarb) [1129-41-5] VOEYXMAFNDNNED-UHFFFAOYSA-N | $1.2\times10^4$ | | HSDB (2015) | V | |
| phenylcarbamic acid, ethyl ester $C_9H_{11}NO_2$ [101-99-5] LBKPGNUOUPTQKA-UHFFFAOYSA-N | $3.4\times10^2$ | | HSDB (2015) | Q | 99 |
| ethyl anthranilate $C_9H_{11}NO_2$ [87-25-2] TWLLPUMZVVGILS-UHFFFAOYSA-N | $6.2\times10^2$ | | HSDB (2015) | Q | 99 |
| benzocaine $C_9H_{11}NO_2$ [94-09-7] BLFLLBZGZJTVJG-UHFFFAOYSA-N | $6.2\times10^2$ | | HSDB (2015) | Q | 99 |
| 1-(4-aminophenyl)-1-propanone $C_9H_{11}NO$ (4-aminopropiophenone) [70-69-9] FSWXOANXOQPCFF-UHFFFAOYSA-N | $2.1\times10^3$ | | HSDB (2015) | Q | 99 |
| (4-ethoxyphenyl)urea $C_9H_{12}N_2O_2$ (dulcin) [150-69-6] GGLIEWRLXDLBBF-UHFFFAOYSA-N | $6.2\times10^5$ | | HSDB (2015) | Q | 447 |
| fenuron $C_9H_{12}N_2O$ [101-42-8] XXOYNJXVWVNOOJ-UHFFFAOYSA-N | $4.9\times10^3$ $8.7\times10^2$ $3.7\times10^3$ $3.6\times10^1$ $9.6$ $1.0\times10^4$ $4.4$ | | Duchowicz et al. (2020) Mackay et al. (2006d) Suntio et al. (1988) Barcelo and Hennion (1997) Duchowicz et al. (2020) HSDB (2015) Goodarzi et al. (2010) | V V V X Q Q Q | 186  12 567  99 568, 571 |
| epinephrine $C_9H_{13}NO_3$ [51-43-4] UCTWMZQNUQWSLP-UHFFFAOYSA-N | $1.4\times10^{13}$ | | HSDB (2015) | Q | 99 |
| meprobamate $C_9H_{18}N_2O_4$ [57-53-4] NPPQSCRMBWNHMW-UHFFFAOYSA-N | $5.5\times10^4$ | | HSDB (2015) | Q | 99 |



Table A4.4: Amines, amides, amino acids (C, H, O, N) (... continued)

| Substance<br>Formula<br>(Trivial Name)<br>[CAS Registry Number]<br>InChIKey | $H_s^{cp}$<br>(at $T^{\ominus}$)<br><br>$\left[\dfrac{\mathrm{mol}}{\mathrm{m^3\,Pa}}\right]$ | $\dfrac{\mathrm{d}\ln H_s^{cp}}{\mathrm{d}(1/T)}$<br><br>[K] | Reference | Type | Note |
|---|---|---|---|---|---|
| propamocarb<br>$C_9H_{20}N_2O_2$<br>[24579-73-5]<br>WZZLDXDUQPOXNW-UHFFFAOYSA-N | $2.1\times10^3$<br>$1.6\times10^3$<br>$1.6\times10^3$<br>$3.3\times10^2$<br>$6.7\times10^3$<br>$1.2\times10^8$ | | Keshavarz et al. (2022)<br>Duchowicz et al. (2020)<br>Hilal et al. (2008)<br>Modarresi et al. (2007)<br>Duchowicz et al. (2020)<br>Maniere et al. (2011) | Q<br>Q<br>Q<br>Q<br>?<br>? | <br>184<br><br>67<br>185, 21<br>241, 165 |
| proximpham<br>$C_{10}H_{12}N_2O_2$<br>[2828-42-4]<br>LATYTXGNKNKTDS-UHFFFAOYSA-N | $3.9\times10^3$ | | MacBean (2012a) | ? | 12 |
| dioxacarb<br>$C_{10}H_{13}NO_4$<br>[6988-21-2]<br>SDKQRNRRDYRQKY-UHFFFAOYSA-N | $6.7\times10^5$<br>$2.2\times10^4$<br>$6.7\times10^5$ | | Duchowicz et al. (2020)<br>Duchowicz et al. (2020)<br>MacBean (2012a) | V<br>Q<br>? | 186 |
| phenol, 3,5-dimethyl-,<br>methylcarbamate<br>$C_{10}H_{13}NO_2$<br>(3,5-xylyl methylcarbamate)<br>[2655-14-3]<br>CVQODEWAPZVVBU-UHFFFAOYSA-N | $5.5\times10^1$<br><br>9.1<br>$2.5\times10^2$<br>$4.3\times10^2$ | | Duchowicz et al. (2020)<br><br>Duchowicz et al. (2020)<br>HSDB (2015)<br>MacBean (2012a) | V<br><br>Q<br>Q<br>? | 186<br><br><br>99 |
| phenylcarbamic acid,<br>1-methylethyl ester<br>$C_{10}H_{13}NO_2$<br>[122-42-9]<br>VXPLXMJHHKHSOA-UHFFFAOYSA-N | $5.4\times10^1$<br><br>$5.5\times10^1$<br>$3.2\times10^{-2}$ | | Duchowicz et al. (2020)<br><br>HSDB (2015)<br>Duchowicz et al. (2020) | V<br><br>V<br>Q | 186 |
| xylylcarb<br>$C_{10}H_{13}NO_2$<br>[2425-10-7]<br>WCJYTPVNMWIZCG-UHFFFAOYSA-N | $9.1\times10^1$<br>$9.4\times10^1$<br>9.1 | | Watanabe (1993)<br>Duchowicz et al. (2020)<br>Duchowicz et al. (2020) | M<br>V<br>Q | <br>186 |
| phenacetin<br>$C_{10}H_{13}NO_2$<br>[62-44-2]<br>CPJSUEIXXCENMM-UHFFFAOYSA-N | $4.6\times10^4$<br>$4.7\times10^4$<br>2.7 | | Duchowicz et al. (2020)<br>HSDB (2015)<br>Duchowicz et al. (2020) | V<br>V<br>Q | 186 |
| ephedrine<br>$C_{10}H_{15}NO$<br>[299-42-3]<br>KWGRBVOPPLSCSI-PSASIEDQSA-N | $1.1\times10^5$ | | HSDB (2015) | Q | 447 |



Table A4.4: Amines, amides, amino acids (C, H, O, N) (... continued)

| Substance<br>Formula<br>(Trivial Name)<br>[CAS Registry Number]<br>InChIKey | $H_s^{cp}$<br>(at $T^\ominus$)<br>$\left[\dfrac{\text{mol}}{\text{m}^3\,\text{Pa}}\right]$ | $\dfrac{\text{d}\ln H_s^{cp}}{\text{d}(1/T)}$<br><br>[K] | Reference | Type | Note |
|---|---|---|---|---|---|
| $m$-cumenyl methylcarbamate<br>$C_{11}H_{15}NO_2$<br>(3-isopropylphenyl methyl<br>carbamate)<br>[64-00-6]<br>GYKXQTKSWLAUIT-UHFFFAOYSA-N | $1.6 \times 10^2$ | | HSDB (2015) | Q | 99 |
| propoxur<br>$C_{11}H_{15}NO_3$<br>[114-26-1]<br>ISRUGXGCCGIOQO-UHFFFAOYSA-N | $2.9 \times 10^3$<br>$5.1 \times 10^5$<br>$7.1 \times 10^3$<br>$7.7$<br>$7.6 \times 10^{-2}$<br>$1.4$ | | HSDB (2015)<br>Mackay et al. (2006d)<br>Siebers et al. (1994)<br>Suntio et al. (1988)<br>Barcelo and Hennion (1997)<br>Goodarzi et al. (2010) | V<br>V<br>V<br>V<br>X<br>Q | <br><br><br>12<br>567<br>568 |
| methocarbamol<br>$C_{11}H_{15}NO_5$<br>[532-03-6]<br>GNXFOGHNGIVQEH-UHFFFAOYSA-N | $1.5 \times 10^{10}$ | | HSDB (2015) | Q | 99 |
| aminocarb<br>$C_{11}H_{16}N_2O_2$<br>[2032-59-9]<br>IMIDOCRTMDIQIJ-UHFFFAOYSA-N | $1.9 \times 10^3$<br>$1.8 \times 10^4$ | | Mackay et al. (2006d)<br>HSDB (2015) | V<br>Q | <br>99 |
| monodesmethylisoproturon<br>$C_{11}H_{16}N_2O$<br>[34123-57-4]<br>DOULWWSSZVEPIN-UHFFFAOYSA-N | $2.8 \times 10^5$ | | Otto et al. (1997) | V | |
| cycluron<br>$C_{11}H_{22}N_2O$<br>[2163-69-1]<br>DQZCVNGCTZLGAQ-UHFFFAOYSA-N | $8.2 \times 10^2$ | | HSDB (2015) | Q | 99 |
| methylneodecanamide<br>$C_{11}H_{23}NO$<br>[105726-67-8]<br>GELCOLZWXWHMIB-UHFFFAOYSA-N | $4.1 \times 10^1$ | | HSDB (2015) | Q | 99 |
| isoprocarb<br>$C_{11}H_{15}NO_2$<br>[2631-40-5]<br>QBSJMKIUCUGGNG-UHFFFAOYSA-N | $7.4 \times 10^2$<br>$7.6$ | | Duchowicz et al. (2020)<br>Duchowicz et al. (2020) | V<br>Q | 186 |
| 4-(phenylazo)phenol<br>$C_{12}H_{10}N_2O$<br>(4-hydroxyazobenzene)<br>[1689-82-3]<br>BEYOBVMPDRKTNR-UHFFFAOYSA-N | $1.5 \times 10^4$<br>$1.5 \times 10^4$<br>$1.0 \times 10^4$ | | Duchowicz et al. (2020)<br>HSDB (2015)<br>Duchowicz et al. (2020) | V<br>V<br>Q | 186 |



Table A4.4: Amines, amides, amino acids (C, H, O, N) (...continued)

| Substance Formula (Trivial Name) [CAS Registry Number] InChIKey | $H_s^{cp}$ (at $T^{\ominus}$) $\left[\dfrac{\mathrm{mol}}{\mathrm{m^3\,Pa}}\right]$ | $\dfrac{\mathrm{d}\ln H_s^{cp}}{\mathrm{d}(1/T)}$ [K] | Reference | Type | Note |
|---|---|---|---|---|---|
| 1-naphthaleneacetamide $C_{12}H_{11}NO$ [86-86-2] XFNJVKMNNVCYEK-UHFFFAOYSA-N | $9.8\times10^7$ | | Maniere et al. (2011) | ? | 12, 165 |
| carbetamide $C_{12}H_{16}N_2O_3$ [16118-49-3] AMRQXHFXNZFDCH-UHFFFAOYSA-N | 1.1 3.7 $5.2\times10^7$ | | Barcelo and Hennion (1997) Goodarzi et al. (2010) Maniere et al. (2011) | X Q ? | 567 568 12, 165 |
| N-nitrosodiphenylamine $C_{12}H_{10}N_2O$ (N,N-Diphenylnitrosamine) [86-30-6] UBUCNCOMADRQHX-UHFFFAOYSA-N | $8.7\times10^{-3}$ $8.7\times10^{-3}$ $1.5\times10^{-2}$ 8.2 | | Mackay et al. (2006d) Mackay et al. (1995) Mackay et al. (1995) HSDB (2015) | V V C Q | 99 |
| carbaryl $C_{12}H_{11}NO_2$ [63-25-2] CVXBEEMKQHEXEN-UHFFFAOYSA-N | $>9.9\times10^1$ $3.6\times10^3$ $3.0\times10^3$ $2.2\times10^4$ $2.3\times10^3$ $7.7\times10^2$ $7.6\times10^1$ $2.3\times10^3$ $3.5\times10^3$ $1.2\times10^2$ 8.1 $1.4\times10^3$ $3.1\times10^3$ | | Mabury and Crosby (1996) Watanabe (1993) Duchowicz et al. (2020) Mackay et al. (2006d) Meylan and Howard (1991) Suntio et al. (1988) Barcelo and Hennion (1997) Howard and Meylan (1997) Armbrust (2000) Duchowicz et al. (2020) Goodarzi et al. (2010) Hilal et al. (2008) Meylan and Howard (1991) | M M V V V V X X C Q Q Q Q | 186 12 567 446 568, 571 |
| 4,4'-oxybisbenzenamine $C_{12}H_{12}N_2O$ (bis(4-aminophenyl) ether) [101-80-4] HLBLWEWZXPIGSM-UHFFFAOYSA-N | $6.6\times10^5$ | | HSDB (2015) | Q | 99 |
| carbofuran $C_{12}H_{15}NO_3$ [1563-66-2] DUEPRVBVGDRKAG-UHFFFAOYSA-N | $2.4\times10^1$ $>9.9\times10^1$ $2.2\times10^4$ $2.0\times10^4$ $2.0\times10^3$ $1.9\times10^1$ $4.6\times10^1$ | | Chao et al. (2017) Mabury and Crosby (1996) HSDB (2015) Mackay et al. (2006d) Suntio et al. (1988) Barcelo and Hennion (1997) Goodarzi et al. (2010) | M M V V V X Q | 12 567 568 |





Table A4.4: Amines, amides, amino acids (C, H, O, N) (... continued)

| Substance Formula (Trivial Name) [CAS Registry Number] InChIKey | $H_s^{cp}$ (at $T^{\ominus}$) $\left[\dfrac{\text{mol}}{\text{m}^3\,\text{Pa}}\right]$ | $\dfrac{\mathrm{d}\ln H_s^{cp}}{\mathrm{d}(1/T)}$ [K] | Reference | Type | Note |
|---|---|---|---|---|---|
| phenol, 3-methyl-5-(1-methylethyl)-, methylcarbamate | $1.1\times10^2$ | | Duchowicz et al. (2020) | V | 186 |
| $C_{12}H_{16}NO_2$ | $1.1\times10^2$ | | HSDB (2015) | V | |
| (promecarb) | 4.1 | | Duchowicz et al. (2020) | Q | |
| [2631-37-0] | $3.1\times10^2$ | | MacBean (2012a) | ? | |
| DTAPQAJKAFRNJB-UHFFFAOYSA-N | | | | | |
| fenobucarb | $1.5\times10^2$ | | Watanabe (1993) | M | |
| $C_{12}H_{17}NO_2$ | $1.7\times10^2$ | | Duchowicz et al. (2020) | V | 186 |
| [3766-81-2] | 8.2 | | Duchowicz et al. (2020) | Q | |
| DIRFUJHNVNOBMY-UHFFFAOYSA-N | | | | | |
| diethyltoluamide | $4.7\times10^2$ | | HSDB (2015) | Q | 99 |
| $C_{12}H_{17}NO$ | | | | | |
| (DEET) | | | | | |
| [134-62-3] | | | | | |
| MMOXZBCLCQITDF-UHFFFAOYSA-N | | | | | |
| N,N-dimethyl-N'-[4-(1-methylethyl)phenyl]-urea | $8.8\times10^4$ | | Duchowicz et al. (2020) | V | 186 |
| $C_{12}H_{18}N_2O$ | $8.1\times10^4$ | | Mackay et al. (2006d) | V | |
| (isoproturon) | $9.5\times10^4$ | | Otto et al. (1997) | V | |
| [34123-59-6] | $1.1\times10^5$ | | Siebers et al. (1994) | V | |
| PUIYMUZLKQOUOZ-UHFFFAOYSA-N | 2.3 | | Duchowicz et al. (2020) | Q | |
| dimorpholinodiethyl ether | $1.8\times10^3$ | 2600 | Nguyen (2013) | M | 11 |
| $C_{12}H_{24}N_2O_3$ | | | | | |
| [6425-39-4] | | | | | |
| ZMSQJSMSLXVTKN-UHFFFAOYSA-N | | | | | |
| carisoprodol | $1.4\times10^4$ | | HSDB (2015) | Q | 99 |
| $C_{12}H_{24}N_2O_4$ | | | | | |
| [78-44-4] | | | | | |
| OFZCIYFFPZCNJE-UHFFFAOYSA-N | | | | | |
| 4-hydroxy-4'-nitroazobenzene | $4.0\times10^5$ | | Duchowicz et al. (2020) | V | 186 |
| $C_{12}H_9N_3O_3$ | $1.4\times10^6$ | | Duchowicz et al. (2020) | Q | |
| [1435-60-5] | | | | | |
| NRJPVIOTANUINF-YPKPFQOOSA-N | | | | | |
| 2,4-dinitro-N-phenyl-benzenamine | $1.3\times10^4$ | | Duchowicz et al. (2020) | V | 186 |
| $C_{12}H_9N_3O_4$ | $6.6\times10^5$ | | Duchowicz et al. (2020) | Q | |
| [961-68-2] | | | | | |
| RHTVQEPJVKUMPI-UHFFFAOYSA-N | | | | | |
| N-(2,4-dinitrophenyl)-N-(4-hydroxyphenyl)amine | $6.8\times10^6$ | | Duchowicz et al. (2020) | V | 186 |
| $C_{12}H_9N_3O_5$ | $1.4\times10^9$ | | Duchowicz et al. (2020) | Q | |
| [119-15-3] | | | | | |
| BCPQALWAROJVLE-UHFFFAOYSA-N | | | | | |



Table A4.4: Amines, amides, amino acids (C, H, O, N) (...continued)

| Substance Formula (Trivial Name) [CAS Registry Number] InChIKey | $H_s^{cp}$ (at $T^\ominus$) $\left[\dfrac{\text{mol}}{\text{m}^3\,\text{Pa}}\right]$ | $\dfrac{\text{d}\ln H_s^{cp}}{\text{d}(1/T)}$ [K] | Reference | Type | Note |
|---|---|---|---|---|---|
| disperse orange 3 $C_{12}H_{10}N_4O_2$ [730-40-5] UNBOSJFEZZJZLR-UHFFFAOYSA-N | $3.5\times10^4$ $1.9\times10^5$ | | Duchowicz et al. (2020) Duchowicz et al. (2020) | V Q | 186 |
| diethylpropion $C_{13}H_{19}NO$ [90-84-6] XXEPPPIWZFICOJ-UHFFFAOYSA-N | $4.3\times10^1$ | | HSDB (2015) | Q | 99 |
| salbutamol $C_{13}H_{21}NO_3$ (albuterol) [18559-94-9] NDAUXUAQIAJITI-UHFFFAOYSA-N | $1.5\times10^{10}$ | | HSDB (2015) | Q | 99 |
| disperse blue 1 $C_{14}H_{12}N_4O_2$ [2475-45-8] JSFUMBWFPQSADC-UHFFFAOYSA-N | $4.7\times10^1$ | | HSDB (2015) | V | |
| 3,3'-dimethoxybenzidine $C_{14}H_{16}N_2O_2$ [119-90-4] JRBJSXQPQWSCCF-UHFFFAOYSA-N | $2.1\times10^5$ | | HSDB (2015) | Q | 99 |
| aspartame $C_{14}H_{18}N_2O_5$ [22839-47-0] IAOZJIPTCAWIRG-QWRGUYRKSA-N | $3.9\times10^{12}$ | | HSDB (2015) | Q | 99 |
| dinobuton $C_{14}H_{18}N_2O_7$ (dessin) [973-21-7] HDWLUGYOLUHEMN-UHFFFAOYSA-N | $6.2\times10^2$ | | HSDB (2015) | Q | 99 |
| N-(2-methylcyclohexyl)-N'-phenylurea $C_{14}H_{20}N_2O$ (siduron) [1982-49-6] JXVIIQLNUPXOII-UHFFFAOYSA-N | $1.5\times10^5$ | | HSDB (2015) | V | |
| butralin $C_{14}H_{21}N_3O_4$ [33629-47-9] SPNQRCTZKIBOAX-UHFFFAOYSA-N | 2.0 2.0 | | HSDB (2015) Mackay et al. (2006d) | V V | |



Table A4.4: Amines, amides, amino acids (C, H, O, N) (. . . continued)

| Substance Formula (Trivial Name) [CAS Registry Number] InChIKey | $H_s^{cp}$ (at $T^{\ominus}$) $\left[\dfrac{\text{mol}}{\text{m}^3\,\text{Pa}}\right]$ | $\dfrac{\text{d}\ln H_s^{cp}}{\text{d}(1/T)}$ [K] | Reference | Type | Note |
|---|---|---|---|---|---|
| lauramine oxide $C_{14}H_{31}NO$ [1643-20-5] SYELZBGXAIXKHU-UHFFFAOYSA-N | $1.5\times10^5$ | | HSDB (2015) | Q | 99 |
| 2-aminoanthraquinone $C_{14}H_9NO_2$ [117-79-3] XOGPDSATLSAZEK-UHFFFAOYSA-N | $1.1\times10^5$ $1.1\times10^5$ $3.1\times10^5$ | | Duchowicz et al. (2020) HSDB (2015) Duchowicz et al. (2020) | V V Q | 186 |
| 1-amino-9,10-anthracenedione $C_{14}H_9NO_2$ [82-45-1] KHUFHLFHOQVFGB-UHFFFAOYSA-N | $2.0\times10^3$ $9.8\times10^4$ | | Duchowicz et al. (2020) Duchowicz et al. (2020) | V Q | 186 |
| 1-amino-4-hydroxy-9,10-anthracenedione $C_{14}H_9NO_3$ [116-85-8] AQXYVFBSOOBBQV-UHFFFAOYSA-N | $3.8\times10^5$ $4.2\times10^5$ | | Duchowicz et al. (2020) Duchowicz et al. (2020) | V Q | 186 |
| 1,4-diamino-9,10-anthracenedione $C_{14}H_{10}N_2O_2$ [128-95-0] FBMQNRKSAWNXBT-UHFFFAOYSA-N | $2.1\times10^4$ $1.3\times10^7$ | | Duchowicz et al. (2020) Duchowicz et al. (2020) | V Q | 186 |
| N,N-dimethyl-4-[(4-nitrophenyl)azo]-benzenamine $C_{14}H_{14}N_4O_2$ [2491-74-9] LSFRFLVWCKLQTO-UHFFFAOYSA-N | $1.5\times10^2$ $1.0\times10^4$ | | Duchowicz et al. (2020) Duchowicz et al. (2020) | V Q | 186 |
| karbutilate $C_{14}H_{21}N_3O_3$ [4849-32-5] OWNAXTAAAQTBSP-UHFFFAOYSA-N | $1.8\times10^8$ | | Ebert et al. (2023) | ? | 318 |
| diethofencarb $C_{14}H_{21}NO_4$ [87130-20-9] LNJNFVJKDJYTEU-UHFFFAOYSA-N | $1.2\times10^1$ $1.3\times10^{-1}$ | | Duchowicz et al. (2020) Duchowicz et al. (2020) | V Q | 186 |
| 1-amino-2-methyl-9,10-anthracenedione $C_{15}H_{11}NO_2$ (1-amino-2-methylanthraquinone) [82-28-0] ZLCUIOWQYBYEBG-UHFFFAOYSA-N | $8.2\times10^6$ | | HSDB (2015) | Q | 99 |





Table A4.4: Amines, amides, amino acids (C, H, O, N) (...continued)

| Substance Formula (Trivial Name) [CAS Registry Number] InChIKey | $H_s^{cp}$ (at $T^\ominus$) $\left[\dfrac{\text{mol}}{\text{m}^3\,\text{Pa}}\right]$ | $\dfrac{\text{d}\ln H_s^{cp}}{\text{d}(1/T)}$ [K] | Reference | Type | Note |
|---|---|---|---|---|---|
| N-2-fluorenylacetamide $C_{15}H_{13}NO$ (2-acetylaminofluorene) [53-96-3] CZIHNRWJTSTCEX-UHFFFAOYSA-N | $5.2\times10^4$ | | HSDB (2015) | Q | 99 |
| tebutam $C_{15}H_{23}NO$ [35256-85-0] RJKCKKDSSSRYCB-UHFFFAOYSA-N | $3.8\times10^1$ $1.5\times10^{-1}$ $6.7\times10^1$ | | Duchowicz et al. (2020) Duchowicz et al. (2020) MacBean (2012a) | V Q ? | 186 |
| isopropalin $C_{15}H_{23}N_3O_4$ [33820-53-0] NEKOXWSIMFDGMA-UHFFFAOYSA-N | $1.9\times10^{-1}$ | | Mackay et al. (2006d) | V | |
| dodine $C_{15}H_{33}N_3O_2$ (doguadine) [2439-10-3] YIKWKLYQRFRGPM-UHFFFAOYSA-N | $1.1\times10^5$ $9.9\times10^5$ $1.1\times10^5$ | | Duchowicz et al. (2020) Duchowicz et al. (2020) Maniere et al. (2011) | V Q ? | 186 241, 165 |
| metalaxyl $C_{15}H_{21}NO_4$ [57837-19-1] ZQEIXNIJLIKNTD-UHFFFAOYSA-N | $3.3\times10^3$ $4.0\times10^4$ $8.5\times10^4$ | | HSDB (2015) Mackay et al. (2006d) Burkhard and Guth (1981) | V V V | |
| metalaxyl-m $C_{15}H_{21}NO_4$ [70630-17-0] ZQEIXNIJLIKNTD-GFCCVEGCSA-N | $2.8\times10^4$ $2.1\times10^2$ $2.9\times10^4$ | | Duchowicz et al. (2020) Duchowicz et al. (2020) Maniere et al. (2011) | V Q ? | 186 165 |
| metoprolol $C_{15}H_{25}NO_3$ [37350-58-6] IUBSYMUCCVWXPE-UHFFFAOYSA-N | $4.7\times10^5$ | | HSDB (2015) | Q | 99 |
| 1-(methylamino)anthraquinone $C_{15}H_{11}NO_2$ [82-38-2] SVTDYSXXLJYUTM-UHFFFAOYSA-N | $3.3\times10^2$ $7.1\times10^4$ | | Duchowicz et al. (2020) Duchowicz et al. (2020) | V Q | 186 |
| 4'-[(2-hydroxy-5-methylphenyl)azo]acetanilide $C_{15}H_{15}N_3O_2$ [2832-40-8] PXOZAFXVEWKXED-UHFFFAOYSA-N | $6.6\times10^5$ $3.2\times10^5$ | | Duchowicz et al. (2020) Duchowicz et al. (2020) | V Q | 186 |





Table A4.4: Amines, amides, amino acids (C, H, O, N) (...continued)

| Substance Formula (Trivial Name) [CAS Registry Number] InChIKey | $H_s^{cp}$ (at $T^{\ominus}$) $\left[\dfrac{\mathrm{mol}}{\mathrm{m^3\,Pa}}\right]$ | $\dfrac{\mathrm{d}\ln H_s^{cp}}{\mathrm{d}(1/T)}$ [K] | Reference | Type | Note |
|---|---|---|---|---|---|
| desmedipham $C_{16}H_{16}N_2O_4$ [13684-56-5] WZJZMXBKUWKXTQ-UHFFFAOYSA-N | $2.3\times10^6$ | | Ebert et al. (2023) | ? | 365 |
| phenmedipham $C_{16}H_{16}N_2O_4$ (betanal) [13684-63-4] IDOWTHOLJBTAFI-UHFFFAOYSA-N | $1.2\times10^7$ $1.2\times10^7$ $1.2\times10^5$ 4.5 $5.3\times10^2$ $2.0\times10^7$ | | Duchowicz et al. (2020) HSDB (2015) Barcelo and Hennion (1997) Duchowicz et al. (2020) Goodarzi et al. (2010) Maniere et al. (2011) | V V X Q Q ? | 186 567 568, 569 12, 165 |
| fenam $C_{16}H_{17}NO$ [957-51-7] QAHFOPIILNICLA-UHFFFAOYSA-N | $2.7\times10^5$ $4.1\times10^5$ $2.7\times10^5$ $4.9\times10^1$ | | Duchowicz et al. (2020) HSDB (2015) Mackay et al. (2006d) Duchowicz et al. (2020) | V V V Q | 186 |
| difenoxuron $C_{16}H_{18}N_2O_3$ [14214-32-5] AMVYOVYGIJXTQB-UHFFFAOYSA-N | $5.6\times10^7$ | | MacBean (2012a) | ? | |
| butacarb $C_{16}H_{25}NO_2$ [2655-19-8] SLZWBCGZQRRUNG-UHFFFAOYSA-N | $2.2\times10^2$ | | HSDB (2015) | V | |
| oseltamivir $C_{16}H_{28}N_2O_4$ [196618-13-0] VSZGPKBBMSAYNT-RRFJBIMHSA-N | $3.4\times10^{10}$ | | HSDB (2015) | Q | 99 |
| N,N-bis(2-hydroxyethyl)dodecanamide $C_{16}H_{33}NO_3$ [120-40-1] AOMUHOFOVNGZAN-UHFFFAOYSA-N | $4.6\times10^6$ | | HSDB (2015) | Q | 99 |
| 1,4-bis(methylamino)-9,10-anthracenedione $C_{16}H_{14}N_2O_2$ [2475-44-7] QOSTVEDABRQTSU-UHFFFAOYSA-N | $5.2\times10^4$ $7.5\times10^6$ | | Duchowicz et al. (2020) Duchowicz et al. (2020) | V Q | 186 |
| metominostrobin $C_{16}H_{16}N_2O_3$ [133408-50-1] HIIRDDUVRXCDBN-SDXDJHTJSA-N | $2.5\times10^4$ $8.6\times10^6$ | | Duchowicz et al. (2020) Duchowicz et al. (2020) | V Q | 186 |



Table A4.4: Amines, amides, amino acids (C, H, O, N) (...continued)

| Substance<br>Formula<br>(Trivial Name)<br>[CAS Registry Number]<br>InChIKey | $H_s^{cp}$<br>(at $T^\ominus$)<br>$\left[\dfrac{\text{mol}}{\text{m}^3\,\text{Pa}}\right]$ | $\dfrac{\text{d}\ln H_s^{cp}}{\text{d}(1/T)}$<br><br>[K] | Reference | Type | Note |
|---|---|---|---|---|---|
| 1-[(2-methoxyphenyl)azo]-2-naphthol<br>$C_{17}H_{14}N_2O_2$<br>[1229-55-6]<br>ALLOLPOYFRLCCX-VHEBQXMUSA-N | $9.0\times10^4$ | | HSDB (2015) | Q | 99 |
| furalaxyl<br>$C_{17}H_{19}NO_4$<br>[57646-30-7]<br>CIEXPHRYOLIQQD-UHFFFAOYSA-N | $1.1\times10^4$ | | MacBean (2012a) | ? | |
| fenoxycarb<br>$C_{17}H_{19}NO_4$<br>[72490-01-8]<br>HJUFTIJOISQSKQ-UHFFFAOYSA-N | $2.3\times10^4$<br>$2.5\times10^4$<br>$2.3\times10^7$<br>$1.2\times10^4$<br>$8.4\times10^2$<br>$8.4\times10^2$<br>$3.0\times10^4$ | | Duchowicz et al. (2020)<br>Duchowicz et al. (2020)<br>HSDB (2015)<br>Mackay et al. (2006d)<br>Duchowicz et al. (2020)<br>Duchowicz et al. (2020)<br>Maniere et al. (2011) | V<br>V<br>V<br>V<br>Q<br>Q<br>? | 186<br>186<br><br><br><br><br>241, 165 |
| bifenazate<br>$C_{17}H_{20}N_2O_3$<br>[149877-41-8]<br>VHLKTXFWDRXILV-UHFFFAOYSA-N | $1.0\times10^3$<br>$>9.9\times10^2$ | | MacBean (2012b)<br>Maniere et al. (2011) | X<br>? | 350<br>12, 165 |
| napropamide<br>$C_{17}H_{21}NO_2$<br>[15299-99-7]<br>WXZVAROIGSFCFJ-UHFFFAOYSA-N | $1.2\times10^4$<br>$1.2\times10^4$ | | HSDB (2015)<br>Maniere et al. (2011) | V<br>? | <br>241, 165 |
| napropamide-M<br>$C_{17}H_{21}NO_2$<br>[41643-35-0]<br>WXZVAROIGSFCFJ-CYBMUJFWSA-N | $3.8\times10^4$ | | Ebert et al. (2023) | ? | 318 |
| padimate O<br>$C_{17}H_{27}NO_2$<br>[21245-02-3]<br>WYWZRNAHINYAEF-UHFFFAOYSA-N | $2.5$ | | HSDB (2015) | Q | 447 |
| nadolol<br>$C_{17}H_{27}NO_4$<br>[42200-33-9]<br>VWPOSFSPZNDTMJ-UHFFFAOYSA-N | $7.0\times10^8$ | | HSDB (2015) | Q | 99 |
| 2,6-di-*tert*-butyl-4-(dimethylaminomethyl)phenol<br>$C_{17}H_{29}NO$<br>[88-27-7]<br>VMZVBRIIHDRYGK-UHFFFAOYSA-N | $4.8\times10^3$<br>$2.4\times10^2$<br>$1.3$<br>$4.8\times10^1$ | | Zhang et al. (2010)<br>Zhang et al. (2010)<br>Zhang et al. (2010)<br>Zhang et al. (2010) | Q<br>Q<br>Q<br>Q | 287, 288<br>287, 289<br>287, 290<br>287, 291 |



Table A4.4: Amines, amides, amino acids (C, H, O, N) (. . . continued)

| Substance Formula (Trivial Name) [CAS Registry Number] InChIKey | $H_s^{cp}$ (at $T^{\ominus}$) $\left[ \dfrac{\mathrm{mol}}{\mathrm{m^3\,Pa}} \right]$ | $\dfrac{\mathrm{d}\ln H_s^{cp}}{\mathrm{d}(1/T)}$ [K] | Reference | Type | Note |
|---|---|---|---|---|---|
| mepronil $C_{17}H_{19}NO_2$ [55814-41-0] BCTQJXQXJVLSIG-UHFFFAOYSA-N | $8.4\times10^2$ 4.7 | | Duchowicz et al. (2020) Duchowicz et al. (2020) | V Q | 186 |
| daimuron $C_{17}H_{20}N_2O$ [42609-52-9] NNYRZQHKCHEXSD-UHFFFAOYSA-N | $9.9\times10^3$ $2.6\times10^1$ | | Duchowicz et al. (2020) Duchowicz et al. (2020) | V Q | 186 |
| naptalam $C_{18}H_{13}NO_3$ [132-66-1] JXTHEWSKYLZVJC-UHFFFAOYSA-N | $>2.3\times10^{10}$ | | MacBean (2012a) | ? | |
| citrus red 2 $C_{18}H_{16}N_2O_3$ [6358-53-8] ULLJIFAIUSUJBP-ZZEZOPTASA-N | $1.9\times10^7$ | | HSDB (2015) | Q | 99 |
| kresoxim-methyl $C_{18}H_{19}NO_4$ [143390-89-0] ZOTBXTZVPHCKPN-UHFFFAOYSA-N | $2.7\times10^3$ $2.8\times10^3$ | | HSDB (2015) Maniere et al. (2011) | V ? | 241, 165 |
| dinocap $C_{18}H_{24}N_2O_6$ [39300-45-3] RNDNTTTUTQAXII-GORDUTHDSA-N | $2.1\times10^3$ | | HSDB (2015) | V | |
| orysastrobin $C_{18}H_{25}N_5O_5$ [248593-16-0] JHIPUJPTQJYEQK-ZLHHXESBSA-N | $2.9\times10^5$ | | Ebert et al. (2023) | ? | 318 |
| spiroxamine $C_{18}H_{35}NO_2$ [118134-30-8] PUYXTUJWRLOUCW-UHFFFAOYSA-N | $4.0\times10^2$ $2.0\times10^2$ | | Maniere et al. (2011) Maniere et al. (2011) | ? ? | 12, 165 12, 165 |
| capsaicin $C_{18}H_{27}NO_3$ [404-86-4] YKPUWZUDDOIDPM-SOFGYWHQSA-N | $9.9\times10^7$ | | HSDB (2015) | Q | 99 |
| mandestrobin $C_{19}H_{23}NO_3$ [173662-97-0] PDPWCKVFIFAQIQ-UHFFFAOYSA-N | $1.5\times10^6$ | | Maniere et al. (2011) | ? | 12, 165 |



Table A4.4: Amines, amides, amino acids (C, H, O, N) (... continued)

| Substance Formula (Trivial Name) [CAS Registry Number] InChIKey | $H_s^{cp}$ (at $T^{\ominus}$) $\left[\dfrac{\text{mol}}{\text{m}^3\,\text{Pa}}\right]$ | $\dfrac{\mathrm{d}\ln H_s^{cp}}{\mathrm{d}(1/T)}$ [K] | Reference | Type | Note |
|---|---|---|---|---|---|
| dimoxystrobin $C_{19}H_{22}N_2O_3$ [149961-52-4] WXUZAHCNPWONDH-DYTRJAOYSA-N | $2.2{\times}10^7$ | | Maniere et al. (2011) | ? | 241, 165 |
| $(RS)$-$\alpha$-2-naphthoxypropionanilide $C_{19}H_{17}NO_2$ (naproanilide) [52570-16-8] LVKTWOXHRYGDMM-UHFFFAOYSA-N | $1.6{\times}10^5$ $7.5{\times}10^4$ | | Hilal et al. (2008) Modarresi et al. (2007) | Q Q | 67 |
| phenylbutazone $C_{19}H_{20}N_2O_2$ [50-33-9] VYMDGNCVAMGZFE-UHFFFAOYSA-N | $1.5{\times}10^3$ | | HSDB (2015) | Q | 99 |
| phenisopham $C_{19}H_{22}N_2O_4$ [57375-63-0] PWEOEHNGYFXZLI-UHFFFAOYSA-N | $1.3{\times}10^4$ | | MacBean (2012a) | ? | |
| formoterol $C_{19}H_{24}N_2O_4$ [73573-87-2] BPZSYCZIITTYBL-ORAYPTAESA-N | $1.1{\times}10^{17}$ | | HSDB (2015) | Q | 99 |
| benalaxyl-m $C_{20}H_{23}NO_3$ [98243-83-5] CJPQIRJHIZUAQP-MRXNPFEDSA-N | $4.3{\times}10^3$ | | Maniere et al. (2011) | ? | 12, 165 |
| benalaxyl $C_{20}H_{23}NO_3$ [71626-11-4] CJPQIRJHIZUAQP-UHFFFAOYSA-N | $8.5{\times}10^1$ $8.3{\times}10^1$ $5.1{\times}10^1$ $1.5{\times}10^2$ | | Duchowicz et al. (2020) Mackay et al. (2006d) Duchowicz et al. (2020) Maniere et al. (2011) | V V Q ? | 186 12, 165 |
| tralkoxydim $C_{20}H_{27}NO_3$ [87820-88-0] SOTLWPHEAQOHHC-UHFFFAOYSA-N | $4.1{\times}10^4$ | | HSDB (2015) | V | |
| neotame $C_{20}H_{30}N_2O_5$ [165450-17-9] HLIAVLHNDJUHFG-HOTGVXAUSA-N | $4.3{\times}10^3$ | | HSDB (2015) | Q | 99 |
| colchicine $C_{22}H_{25}NO_6$ [64-86-8] IAKHMKGGTNLKSZ-MRXNPFEDSA-N | $5.5{\times}10^{11}$ | | HSDB (2015) | Q | 99 |



Table A4.4: Amines, amides, amino acids (C, H, O, N) (…continued)

| Substance<br>Formula<br>(Trivial Name)<br>[CAS Registry Number]<br>InChIKey | $H_s^{cp}$<br>(at $T^\ominus$)<br>$\left[\dfrac{\text{mol}}{\text{m}^3\,\text{Pa}}\right]$ | $\dfrac{\text{d}\ln H_s^{cp}}{\text{d}(1/T)}$<br><br>[K] | Reference | Type | Note |
|---|---|---|---|---|---|
| tebufenozide<br>$C_{22}H_{28}N_2O_2$<br>[112410-23-8]<br>QYPNKSZPJQQLRK-UHFFFAOYSA-N | $7.6\times10^2$<br>$>1.5\times10^4$ | | HSDB (2015)<br>Maniere et al. (2011) | V<br>? | <br>165 |
| methoxyfenozide<br>$C_{22}H_{28}N_2O_3$<br>[161050-58-4]<br>QCAWEPFNJXQPAN-UHFFFAOYSA-N | $2.6\times10^6$<br>$>6.1\times10^3$ | | HSDB (2015)<br>Maniere et al. (2011) | Q<br>? | 99<br>12, 165 |
| propoxyphene<br>$C_{22}H_{29}NO_2$<br>[469-62-5]<br>XLMALTXPSGQGBX-PGRDOPGGSA-N | $4.3\times10^3$ | | HSDB (2015) | Q | 99 |
| (Z)-13-docosenamide<br>$C_{22}H_{43}NO$<br>(erucamide)<br>[112-84-5]<br>UAUDZVJPLUQNMU-KTKRTIGZSA-N | 3.5 | | HSDB (2015) | Q | 545 |
| butroxydim<br>$C_{24}H_{33}NO_4$<br>[138164-12-2]<br>ZOGDSYNXUXQGHF-XIEYBQDHSA-N | $1.7\times10^4$ | | MacBean (2012a) | ? | |
| 2,2-bis[4-(4-aminophenoxy)phenyl]propane<br>$C_{27}H_{26}N_2O_2$<br>[13080-86-9]<br>KMKWGXGSGPYISJ-UHFFFAOYSA-N | $2.0\times10^8$<br>$2.8\times10^8$<br>$1.0\times10^8$<br>$3.1\times10^{10}$ | | Zhang et al. (2010)<br>Zhang et al. (2010)<br>Zhang et al. (2010)<br>Zhang et al. (2010) | Q<br>Q<br>Q<br>Q | 287, 288<br>287, 289<br>287, 290<br>287, 291 |
| 1,4-bis[(4-methylphenyl)amino]-9,10-anthracenedione<br>$C_{28}H_{22}N_2O_2$<br>(D&C Green No. 6)<br>[128-80-3]<br>TVRGPOFMYCMNRB-UHFFFAOYSA-N | $6.6\times10^{10}$ | | HSDB (2015) | Q | 99 |
| mifepristone<br>$C_{29}H_{35}NO_2$<br>[84371-65-3]<br>VKHAHZOOUSRJNA-GCNJZUOMSA-N | $2.0\times10^{-1}$ | | HSDB (2015) | Q | 99 |
| 2'-anilino-6'-[ethyl(3-methylbutyl)amino]-3'-methylspiro[isobenzofuran-1(3H),9'-[9H]xanthene]-3-one<br>$C_{34}H_{34}N_2O_3$<br>[70516-41-5]<br>HUSIBQLZEMMTCQ-UHFFFAOYSA-N | $8.4\times10^7$<br><br>$2.0\times10^8$<br>$3.5\times10^8$<br>$8.0\times10^8$ | | Zhang et al. (2010)<br><br>Zhang et al. (2010)<br>Zhang et al. (2010)<br>Zhang et al. (2010) | Q<br><br>Q<br>Q<br>Q | 287, 288<br><br>287, 289<br>287, 290<br>287, 291 |





Table A4.4: Amines, amides, amino acids (C, H, O, N) (...continued)

| Substance Formula (Trivial Name) [CAS Registry Number] InChIKey | $H_s^{cp}$ (at $T^\ominus$) $\left[\dfrac{\text{mol}}{\text{m}^3\,\text{Pa}}\right]$ | $\dfrac{\text{d}\ln H_s^{cp}}{\text{d}(1/T)}$ [K] | Reference | Type | Note |
|---|---|---|---|---|---|
| spinetoram | $1.0\times10^2$ | | Maniere et al. (2011) | ? | 241, 570, 165 |
| $C_{42}H_{69}NO_{10}$ | $1.1\times10^4$ | | Maniere et al. (2011) | ? | 241, 570, 165 |
| [935545-74-7] | $1.6\times10^2$ | | Maniere et al. (2011) | ? | 241, 572, 165 |
| GOENIMGKWNZVDA-OAMCMWGQSA-N | $2.0\times10^3$ | | Maniere et al. (2011) | ? | 241, 165 |
| | $2.9\times10^2$ | | Maniere et al. (2011) | ? | 241, 493, 165 |
| | $2.9\times10^3$ | | Maniere et al. (2011) | ? | 241, 493, 165 |
| | $4.3\times10^1$ | | Maniere et al. (2011) | ? | 241, 572, 165 |
| | $2.5\times10^2$ | | Maniere et al. (2011) | ? | 241, 165 |
| spinosad | $5.3\times10^6$ | | Maniere et al. (2011) | ? | 241, 165 |
| $C_{42}H_{71}NO_9$ | $4.3\times10^4$ | | Maniere et al. (2011) | ? | 241, 165 |
| [168316-95-8] | | | | | |
| RQOIAWYOVOXMST-UHFFFAOYSA-N | | | | | |
| emamectin benzoate | $5.8\times10^3$ | | HSDB (2015) | V | |
| $C_{56}H_{81}NO_{15}$ | $2.5\times10^1$ | | Maniere et al. (2011) | ? | 241, 573, 165 |
| [155569-91-8] | $5.9\times10^3$ | | Maniere et al. (2011) | ? | 241, 493, 165 |
| GCKZANITAMOIAR-UHFFFAOYSA-N | $7.7\times10^4$ | | Maniere et al. (2011) | ? | 241, 570, 165 |
| glutamic acid | $9.0\times10^7$ | | Yaws (2003) | X | 237 |
| $C_5H_9NO_4$ | $1.8\times10^{10}$ | | Gharagheizi et al. (2012) | Q | |
| [617-65-2] | $9.6\times10^7$ | | Gharagheizi et al. (2010) | Q | 246 |
| WHUUTDBJXJRKMK-UHFFFAOYSA-N | $9.9\times10^{10}$ | | Saxena and Hildemann (1996) | E | 401 |
| asparagine | $9.9\times10^{10}$ | | Saxena and Hildemann (1996) | E | 401 |
| $C_4H_8N_2O_3$ | | | | | |
| [70-47-3] | | | | | |
| DCXYFEDJOCDNAF-UWTATZPHSA-N | | | | | |
| serine | $3.9\times10^{10}$ | | Saxena and Hildemann (1996) | E | 401 |
| $C_3H_7NO_3$ | | | | | |
| [302-84-1] | | | | | |
| MTCFGRXMJLQNBG-UHFFFAOYSA-N | | | | | |
| glutamine | $3.3\times10^{10}$ | | HSDB (2015) | Q | 447 |
| $C_5H_{10}N_2O_3$ | $9.9\times10^{10}$ | | Saxena and Hildemann (1996) | E | 401 |
| [56-85-9] | | | | | |
| ZDXPYRJPNDTMRX-GSVOUGTGSA-N | | | | | |



Table A4.4: Amines, amides, amino acids (C, H, O, N) (...continued)

| Substance<br>Formula<br>(Trivial Name)<br>[CAS Registry Number]<br>InChIKey | $H_s^{cp}$<br>(at $T^{\ominus}$)<br>$\left[\dfrac{\mathrm{mol}}{\mathrm{m^3\,Pa}}\right]$ | $\dfrac{\mathrm{d}\ln H_s^{cp}}{\mathrm{d}(1/T)}$<br><br>[K] | Reference | Type | Note |
|---|---|---|---|---|---|
| glycine<br>$C_2H_5NO_2$<br>[56-40-6]<br>DHMQDGOQFOQNFH-UHFFFAOYSA-N | $1.2\times10^{11}$<br>$8.9\times10^5$ | 16000 | Brimblecombe et al. (1992)<br>Saxena and Hildemann (1996) | V<br>E | <br>401 |
| arginine<br>$C_6H_{14}N_4O_2$<br>[74-79-3]<br>ODKSFYDXXFIFQN-SCSAIBSYSA-N | $9.9\times10^{14}$ | | Saxena and Hildemann (1996) | E | 401 |
| alanine<br>$C_3H_7NO_2$<br>[302-72-7]<br>QNAYBMKLOCPYGJ-UHFFFAOYSA-N | $3.5\times10^{10}$<br>$5.9\times10^5$ | 16000 | Brimblecombe et al. (1992)<br>Saxena and Hildemann (1996) | V<br>E | <br>401 |
| leucine<br>$C_6H_{13}NO_2$<br>[328-39-2]<br>ROHFNLRQFUQHCH-UHFFFAOYSA-N | $2.0\times10^5$ | | Saxena and Hildemann (1996) | E | 401 |





### A4.5 Heterocycles with oxygen and nitrogen (C, H, O, N)

Table A4.5: Heterocycles with oxygen and nitrogen (C, H, O, N)

| Substance<br>Formula<br>(Trivial Name)<br>[CAS Registry Number]<br>InChIKey | $H_s^{cp}$<br>(at $T^{\ominus}$)<br>$\left[\dfrac{\mathrm{mol}}{\mathrm{m^3\,Pa}}\right]$ | $\dfrac{\mathrm{d}\ln H_s^{cp}}{\mathrm{d}(1/T)}$<br><br>[K] | Reference | Type | Note |
|---|---|---|---|---|---|
| cyanuric acid<br>$C_3H_3N_3O_3$<br>[108-80-5]<br>ZFSLODLOARCGLH-UHFFFAOYSA-N | $1.1\times10^9$<br>$1.1\times10^9$<br>$3.4\times10^5$<br>$4.2\times10^{10}$<br>$4.0\times10^7$ | | HSDB (2015)<br>Zhang et al. (2010)<br>Zhang et al. (2010)<br>Zhang et al. (2010)<br>Zhang et al. (2010) | Q<br>Q<br>Q<br>Q<br>Q | 99<br>287, 288<br>287, 289<br>287, 290<br>287, 291 |
| isoxazole<br>$C_3H_3NO$<br>[288-14-2]<br>CTAPFRYPJLPFDF-UHFFFAOYSA-N | $4.0\times10^{-1}$<br>$5.4\times10^{-1}$<br>$2.4\times10^{-1}$<br>$2.1\times10^{-1}$ | | Duchowicz et al. (2020)<br>Duchowicz et al. (2020)<br>Hilal et al. (2008)<br>Modarresi et al. (2007) | V<br>Q<br>Q<br>Q | 186<br><br><br>67 |
| glycidamide<br>$C_3H_5NO_2$<br>[5694-00-8]<br>FMAZQSYXRGRESX-UHFFFAOYSA-N | $7.7\times10^4$ | | HSDB (2015) | Q | 99 |
| cyclonite<br>$C_3H_6N_6O_6$<br>[121-82-4]<br>XTFIVUDBNACUBN-UHFFFAOYSA-N | $4.9\times10^5$ | | HSDB (2015) | V | |
| 5-methyl-3-(2H)-isoxazolone<br>$C_4H_5NO_2$<br>(hymexazol)<br>[10004-44-1]<br>KGVPNLBXJKTABS-UHFFFAOYSA-N | $4.7\times10^3$<br>$4.7\times10^2$<br>$5.0\times10^3$<br>$1.6\times10^2$<br>$7.1\times10^3$ | | Duchowicz et al. (2020)<br>Duchowicz et al. (2020)<br>Hilal et al. (2008)<br>Modarresi et al. (2007)<br>Maniere et al. (2011) | V<br>Q<br>Q<br>Q<br>? | 186<br><br><br>67<br>241, 165 |
| 3-amino-1H-pyridazin-6-one<br>$C_4H_5N_3O$<br>(maleic hydrazide)<br>[10071-13-3]<br>MMZLICVOTDAZOX-UHFFFAOYSA-N | $1.2\times10^7$ | | Ebert et al. (2023) | ? | 318 |
| maleic hydrazide<br>$C_4H_4N_2O_2$<br>[123-33-1]<br>BGRDGMRNKXEXQD-UHFFFAOYSA-N | $2.5\times10^7$ | | Maniere et al. (2011) | ? | 12, 165 |
| allantoin<br>$C_4H_6N_4O_3$<br>[97-59-6]<br>POJWUDADGALRAB-UHFFFAOYSA-N | $2.9\times10^{12}$ | | HSDB (2015) | Q | 99 |
| 2-pyrrolidinone<br>$C_4H_7NO$<br>[616-45-5]<br>HNJBEVLQSNELDL-UHFFFAOYSA-N | $9.3\times10^3$<br>$9.3\times10^3$<br>$1.2\times10^3$ | | Duchowicz et al. (2020)<br>HSDB (2015)<br>Duchowicz et al. (2020) | V<br>V<br>Q | 186 |



Table A4.5: Heterocycles with oxygen and nitrogen (C, H, O, N) (...continued)

| Substance<br>Formula<br>(Trivial Name)<br>[CAS Registry Number]<br>InChIKey | $H_s^{cp}$<br>(at $T^{\ominus}$)<br>$\left[\dfrac{\text{mol}}{\text{m}^3\,\text{Pa}}\right]$ | $\dfrac{\text{d}\ln H_s^{cp}}{\text{d}(1/T)}$<br><br>[K] | Reference | Type | Note |
|---|---|---|---|---|---|
| 4-nitrosomorpholine<br>$C_4H_8N_2O_2$<br>[59-89-2]<br>ZKXDGKXYMTYWTB-UHFFFAOYSA-N | $3.9\times10^2$<br>$9.0\times10^2$<br>$1.5\times10^2$ | | Mirvish et al. (1976)<br>Hilal et al. (2008)<br>Modarresi et al. (2007) | M<br>Q<br>Q | 14<br><br>67 |
| N-nitrosopyrrolidine<br>$C_4H_8N_2O$<br>[930-55-2]<br>WNYADZVDBIBLJJ-UHFFFAOYSA-N | $1.5\times10^2$<br>$1.9\times10^2$<br>$3.4\times10^1$ | 8500 | Klein (1982)<br>Mirvish et al. (1976)<br>Hilal et al. (2008) | M<br>M<br>Q | <br>14 |
| cyclotetramethylenetetranitramine<br>$C_4H_8N_8O_8$<br>[2691-41-0]<br>UZGLIIJVICEWHF-UHFFFAOYSA-N | $1.1\times10^4$ | | HSDB (2015) | Q | 99 |
| 1-oxa-4-azacyclohexane<br>$C_4H_9NO$<br>(morpholine)<br>[110-91-8]<br>YNAVUWVOSKDBBP-UHFFFAOYSA-N | $1.9\times10^2$<br>8.5<br>8.2<br>$7.3\times10^1$<br>$2.2\times10^1$<br>$1.6\times10^2$<br>$8.1\times10^1$<br>$9.5\times10^1$<br>$1.0\times10^1$<br>$4.2\times10^1$ | 7800<br><br><br>8400 | Nguyen (2013)<br>Duchowicz et al. (2020)<br>HSDB (2015)<br>Cabani et al. (1975a)<br>Duchowicz et al. (2020)<br>Hilal et al. (2008)<br>Modarresi et al. (2007)<br>English and Carroll (2001)<br>Nirmalakhandan et al. (1997)<br>Yaws (1999) | M<br>V<br>V<br>T<br>Q<br>Q<br>Q<br>Q<br>Q<br>? | 11<br>186<br><br><br><br><br>67<br>230, 274<br><br>21, 12 |
| 1-aziridineethanol<br>$C_4H_9NO$<br>[1072-52-2]<br>VYONOYYDEFODAJ-UHFFFAOYSA-N | $1.3\times10^4$ | | HSDB (2015) | Q | 99 |
| 2-ethyl-3-methoxypyrazine<br>$C_4N_2H_3(C_2H_5)OCH_3$<br>[25680-58-4]<br>DPCILIMHENXHQX-UHFFFAOYSA-N | $6.7\times10^{-1}$<br>$2.5\times10^1$<br>2.6 | | Buttery et al. (1971)<br>Hilal et al. (2008)<br>Modarresi et al. (2007) | M<br>Q<br>Q | <br><br>67 |
| 2-isobutyl-3-methoxypyrazine<br>$C_4N_2H_3(C_4H_9)OCH_3$<br>[24683-00-9]<br>UXFSPRAGHGMRSQ-UHFFFAOYSA-N | $1.7\times10^{-1}$<br>$2.0\times10^{-1}$<br>1.3 | | Karl et al. (2003)<br>Buttery et al. (1971)<br>Modarresi et al. (2007) | M<br>M<br>Q | <br><br>67 |
| N-nitrosopiperidine<br>$C_5H_{10}N_2O$<br>[100-75-4]<br>UWSDONTXWQOZFN-UHFFFAOYSA-N | $1.1\times10^1$<br>$2.9\times10^1$<br>9.6<br>3.4 | | Mirvish et al. (1976)<br>Hilal et al. (2008)<br>Modarresi et al. (2007)<br>Katritzky et al. (1998) | M<br>Q<br>Q<br>Q | 14<br><br>67<br> |
| butyl carbamate<br>$C_5H_{11}NO_2$<br>[592-35-8]<br>SKKTUOZKZKCGTB-UHFFFAOYSA-N | $1.1\times10^2$ | | HSDB (2015) | Q | 99 |



Table A4.5: Heterocycles with oxygen and nitrogen (C, H, O, N) (...continued)

| Substance<br>Formula<br>(Trivial Name)<br>[CAS Registry Number]<br>InChIKey | $H_s^{cp}$<br>(at $T^{\ominus}$)<br>$\left[\dfrac{\text{mol}}{\text{m}^3\,\text{Pa}}\right]$ | $\dfrac{\text{d}\ln H_s^{cp}}{\text{d}(1/T)}$<br><br>[K] | Reference | Type | Note |
|---|---|---|---|---|---|
| 4-methyl-1-oxa-4-azacyclohexane | 5.0 | | Du et al. (2017) | M | 478 |
| $C_5H_{11}NO$ | $1.2\times10^1$ | 10000 | Leng et al. (2015a) | M | |
| (N-methylmorpholine; | $1.8\times10^1$ | 8300 | Cabani et al. (1975a) | T | |
| 4-methylmorpholine) | | | | | |
| [109-02-4] | 9.6 | | Du et al. (2017) | Q | 549 |
| SJRJJKPEHAURKC-UHFFFAOYSA-N | 3.6 | | Du et al. (2017) | Q | |
| | 5.7 | | Hilal et al. (2008) | Q | |
| | 6.4 | | Modarresi et al. (2007) | Q | 67 |
| | $1.4\times10^1$ | | English and Carroll (2001) | Q | 230, 231 |
| | $1.7\times10^1$ | | Nirmalakhandan et al. (1997) | Q | |
| allopurinol | $4.9\times10^8$ | | HSDB (2015) | Q | 99 |
| $C_5H_4N_4O$ | | | | | |
| [315-30-0] | | | | | |
| OFCNXPDARWKPPY-UHFFFAOYSA-N | | | | | |
| 4-methoxypyridine | | 7100 | Arnett and Chawla (1979) | ? | 559 |
| $C_5H_4NOCH_3$ | | | | | |
| [620-08-6] | | | | | |
| XQABVLBGNWBWIV-UHFFFAOYSA-N | | | | | |
| N-methyl-2-pyrrolidone | $3.1\times10^3$ | 11000 | Brockbank (2013) | L | 1 |
| $C_5H_9NO$ | $2.1\times10^3$ | 9100 | Bernauer and Dohnal (2009) | M | |
| [872-50-4] | $3.1\times10^3$ | | Kim et al. (2000) | M | |
| SECXISVLQFMRJM-UHFFFAOYSA-N | $2.8\times10^3$ | | Keshavarz et al. (2022) | Q | |
| | $1.0\times10^1$ | | Duchowicz et al. (2020) | Q | 184 |
| | $3.7\times10^1$ | | Modarresi et al. (2007) | Q | 67 |
| | $3.1\times10^3$ | | Duchowicz et al. (2020) | ? | 185, 21 |
| 5,5-dimethyl-2,4- | $3.5\times10^3$ | | HSDB (2015) | Q | 99 |
| imidazolidinedione | | | | | |
| $C_5H_8N_2O_2$ | $3.6\times10^3$ | | Zhang et al. (2010) | Q | 287, 288 |
| [77-71-4] | $1.6\times10^5$ | | Zhang et al. (2010) | Q | 287, 289 |
| YIROYDNZEPTFOL-UHFFFAOYSA-N | $5.1\times10^6$ | | Zhang et al. (2010) | Q | 287, 290 |
| | $1.6\times10^5$ | | Zhang et al. (2010) | Q | 287, 291 |
| 1-methyluracil | $2.3\times10^3$ | | Duchowicz et al. (2020) | V | 186 |
| $C_5H_6N_2O_2$ | $2.1\times10^5$ | | Duchowicz et al. (2020) | Q | |
| [615-77-0] | | | | | |
| XBCXJKGHPABGSD-UHFFFAOYSA-N | | | | | |
| 2-azacycloheptanone | $1.8\times10^5$ | | HSDB (2015) | V | |
| $C_6H_{11}NO$ | $2.0\times10^3$ | | Hwang et al. (1992) | V | |
| (caprolactam) | $3.9\times10^4$ | | Abraham et al. (2019) | Q | |
| [105-60-2] | | | | | |
| JBKVHLHDHHXQEQ-UHFFFAOYSA-N | | | | | |



Table A4.5: Heterocycles with oxygen and nitrogen (C, H, O, N) (...continued)

| Substance Formula (Trivial Name) [CAS Registry Number] InChIKey | $H_s^{cp}$ (at $T^{\ominus}$) $\left[\dfrac{\mathrm{mol}}{\mathrm{m^3\,Pa}}\right]$ | $\dfrac{\mathrm{d}\ln H_s^{cp}}{\mathrm{d}(1/T)}$ [K] | Reference | Type | Note |
|---|---|---|---|---|---|
| N-acetylpyrrolidine $C_6H_{11}NO$ [4030-18-6] LNWWQYYLZVZXKS-UHFFFAOYSA-N | $6.2 \times 10^3$ | | Gibbs et al. (1991) | M | |
| glucosamine $C_6H_{13}NO_5$ [3416-24-8] MSWZFWKMSRAUBD-IVMDWMLBSA-N | $1.3 \times 10^{10}$ | | HSDB (2015) | Q | 99 |
| N-ethylmorpholine $C_6H_{13}NO$ [100-74-3] HVCNXQOWACZAFN-UHFFFAOYSA-N | $4.0 \times 10^2$ | | HSDB (2015) | Q | 99 |
| 3-formylpyridine $C_6H_5NO$ [500-22-1] QJZUKDFHGGYHMC-UHFFFAOYSA-N | $6.5 \times 10^1$ $6.5 \times 10^1$ $1.5 \times 10^1$ $1.0 \times 10^2$ $2.7 \times 10^1$ $6.5 \times 10^1$ $1.2 \times 10^2$ $3.8 \times 10^1$ $6.6 \times 10^1$ | | Abraham et al. (1994a) Keshavarz et al. (2022) Duchowicz et al. (2020) Hilal et al. (2008) Modarresi et al. (2007) Yaffe et al. (2003) English and Carroll (2001) Nirmalakhandan et al. (1997) Duchowicz et al. (2020) | R Q Q Q Q Q Q Q ? | 67 248, 249 230, 274 185, 21 |
| 4-formylpyridine $C_6H_5NO$ [872-85-5] BGUWFUQJCDRPTL-UHFFFAOYSA-N | $5.6 \times 10^1$ $6.5 \times 10^1$ $1.5 \times 10^1$ $1.0 \times 10^2$ $2.2 \times 10^1$ $6.5 \times 10^1$ $6.4 \times 10^1$ $3.8 \times 10^1$ $5.6 \times 10^1$ | | Abraham et al. (1994a) Keshavarz et al. (2022) Duchowicz et al. (2020) Hilal et al. (2008) Modarresi et al. (2007) Yaffe et al. (2003) English and Carroll (2001) Nirmalakhandan et al. (1997) Duchowicz et al. (2020) | R Q Q Q Q Q Q Q ? | 299 67 248, 272 230, 231 185, 21 |
| niacinamide $C_6H_6N_2O$ [98-92-0] DFPAKSUCGFBDDF-UHFFFAOYSA-N | $3.4 \times 10^6$ | | HSDB (2015) | Q | 99 |
| metronidazole $C_6H_9N_3O_3$ [443-48-1] VAOCPAMSLUNLGC-UHFFFAOYSA-N | $5.8 \times 10^5$ | | HSDB (2015) | Q | 99 |
| nicotinic acid $C_6H_5NO_2$ [59-67-6] PVNIIMVLHYAWGP-UHFFFAOYSA-N | $2.5 \times 10^4$ | | Abraham et al. (2019) | Q | |



Table A4.5: Heterocycles with oxygen and nitrogen (C, H, O, N) (...continued)

| Substance Formula (Trivial Name) [CAS Registry Number] InChIKey | $H_s^{cp}$ (at $T^{\ominus}$) $\left[\dfrac{\mathrm{mol}}{\mathrm{m^3\,Pa}}\right]$ | $\dfrac{\mathrm{d}\ln H_s^{cp}}{\mathrm{d}(1/T)}$ [K] | Reference | Type | Note |
|---|---|---|---|---|---|
| isonicotinic acid C$_6$H$_5$NO$_2$ [55-22-1] TWBYWOBDOCUKOW-UHFFFAOYSA-N | $1.6\times10^4$ | | Abraham et al. (2019) | Q | |
| 1-methylthymine C$_6$H$_8$N$_2$O$_2$ [4160-72-9] GKMIDMKPBOUSBQ-UHFFFAOYSA-N | $1.7\times10^4$ | | Ebert et al. (2023) | ? | 365 |
| glydant C$_7$H$_{12}$N$_2$O$_4$ (1,3-dimethylol-5,5-dimethylhydantoin) [6440-58-0] WSDISUOETYTPRL-UHFFFAOYSA-N | $1.4\times10^6$ | | HSDB (2015) | Q | 99 |
| 3-quinuclidinol C$_7$H$_{13}$NO [1619-34-7] IVLICPVPXWEGCA-UHFFFAOYSA-N | $1.7\times10^3$ $1.8\times10^4$ $5.1\times10^3$ $1.2\times10^4$ | | Du et al. (2017) Du et al. (2017) Du et al. (2017) HSDB (2015) | M Q Q Q | 478 549 <br> 99 |
| dinotefuran C$_7$H$_{14}$N$_4$O$_3$ [165252-70-0] YKBZOVFACRVRJN-UHFFFAOYSA-N | $1.5\times10^8$ | | HSDB (2015) | V | |
| 1,2,3-benzotriazin-4(1H)-one C$_7$H$_5$N$_3$O [90-16-4] DMSSTTLDFWKBSX-UHFFFAOYSA-N | $3.1\times10^4$ | | HSDB (2015) | Q | 99 |
| 4-acetylpyridine C$_7$H$_7$NO [1122-54-9] WMQUKDQWMMOHSA-UHFFFAOYSA-N | $1.6\times10^2$ $8.8\times10^1$ $5.0$ $1.9\times10^2$ $4.1\times10^1$ $8.0\times10^1$ $2.7\times10^1$ $1.6\times10^2$ | | Abraham et al. (1994a) Keshavarz et al. (2022) Duchowicz et al. (2020) Hilal et al. (2008) Modarresi et al. (2007) English and Carroll (2001) Nirmalakhandan et al. (1997) Duchowicz et al. (2020) | R Q Q Q Q Q Q ? | <br> <br> 299 <br> 67 230, 231 <br> 185, 21 |
| 3-acetylpyridine C$_7$H$_7$NO [350-03-8] WEGYGNROSJDEIW-UHFFFAOYSA-N | $4.6\times10^2$ $8.8\times10^1$ $5.0$ $1.9\times10^2$ $5.2\times10^1$ $2.7\times10^1$ $4.6\times10^2$ | | Abraham et al. (1994a) Keshavarz et al. (2022) Duchowicz et al. (2020) Hilal et al. (2008) Modarresi et al. (2007) Nirmalakhandan et al. (1997) Duchowicz et al. (2020) | R Q Q Q Q Q ? | <br> <br> 299 <br> 67 <br> 185, 21 |





Table A4.5: Heterocycles with oxygen and nitrogen (C, H, O, N) (... continued)

| Substance Formula (Trivial Name) [CAS Registry Number] InChIKey | $H_s^{cp}$ (at $T^\ominus$) $\left[ \dfrac{\text{mol}}{\text{m}^3\,\text{Pa}} \right]$ | $\dfrac{\text{d} \ln H_s^{cp}}{\text{d}(1/T)}$ [K] | Reference | Type | Note |
|---|---|---|---|---|---|
| theophylline $C_7H_8N_4O_2$ [58-55-9] ZFXYFBGIUFBOJW-UHFFFAOYSA-N | $5.5 \times 10^8$ | | HSDB (2015) | Q | 99 |
| theobromine $C_7H_8N_4O_2$ [83-67-0] YAPQBXQYLJRXSA-UHFFFAOYSA-N | $6.2 \times 10^5$ | | HSDB (2015) | Q | 99 |
| 2-pyridineethanol $C_7H_9NO$ [103-74-2] BXGYBSJAZFGIPX-UHFFFAOYSA-N | $6.6 \times 10^4$ | | HSDB (2015) | Q | 99 |
| caffeine $C_8H_{10}N_4O_2$ [58-08-2] RYYVLZVUVIJVGH-UHFFFAOYSA-N | $9.0 \times 10^5$ | | HSDB (2015) | V | |
| acyclovir $C_8H_{11}N_5O_3$ [59277-89-3] MKUXAQIIEYXACX-UHFFFAOYSA-N | $3.1 \times 10^{16}$ | | HSDB (2015) | Q | 99 |
| 2-methoxy-3-(1-methylethyl)-pyrazine $C_8H_{12}N_2O$ [25773-40-4] NTOPKICPEQUPPH-UHFFFAOYSA-N | $2.1 \times 10^1$ $1.5 \times 10^1$ | | Wu et al. (2022a) Hilal et al. (2008) | Q Q | 413 |
| simeton $C_8H_{15}N_5O$ [673-04-1] HKAMKLBXTLTVCN-UHFFFAOYSA-N | $1.5 \times 10^4$ $2.5 \times 10^4$ | | Hilal et al. (2008) Abraham et al. (2007) | Q Q | |
| N-isobutylmorpholine $C_8H_{17}NO$ [10315-98-7] QKVSMSABRNCNRS-UHFFFAOYSA-N | | 8100 6000 | Kühne et al. (2005) Kühne et al. (2005) | Q ? | |
| phthalimide $C_8H_5NO_2$ [85-41-6] XKJCHHZQLQNZHY-UHFFFAOYSA-N | $9.9 \times 10^2$ | | HSDB (2015) | Q | 99 |
| furazolidone $C_8H_7N_3O_5$ [67-45-8] PLHJDBGFXBMTGZ-UHFFFAOYSA-N | $3.0 \times 10^5$ | | HSDB (2015) | Q | 99 |



Table A4.5: Heterocycles with oxygen and nitrogen (C, H, O, N) (...continued)

| Substance<br>Formula<br>(Trivial Name)<br>[CAS Registry Number]<br>InChIKey | $H_s^{cp}$ (at $T^\ominus$) $\left[\dfrac{\mathrm{mol}}{\mathrm{m^3\,Pa}}\right]$ | $\dfrac{\mathrm{d}\ln H_s^{cp}}{\mathrm{d}(1/T)}$ [K] | Reference | Type | Note |
|---|---|---|---|---|---|
| 1,2,3,6-tetrahydrophthalimide<br>$C_8H_9NO_2$<br>[85-40-5]<br>CIFFBTOJCKSRJY-UHFFFAOYSA-N | $3.3\times10^2$ | | HSDB (2015) | Q | 99 |
| N-nitrosonornicotine<br>$C_9H_{11}N_3O$<br>[16543-55-8]<br>XKABJYQDMJTNGQ-UHFFFAOYSA-N | $5.8\times10^4$ | | HSDB (2015) | Q | 447 |
| 9-[(1,3-dihydroxy-2-propoxy)methyl]guanine<br>$C_9H_{13}N_5O_4$<br>(ganciclovir)<br>[82410-32-0]<br>IRSCQMHQWWYFCW-UHFFFAOYSA-N | $6.6\times10^{17}$ | | HSDB (2015) | Q | 99 |
| 2-*sec*-butyl-3-methoxypyrazine<br>$C_9H_{14}N_2O$<br>[24168-70-5]<br>QMQDJVIJVPEQHE-UHFFFAOYSA-N | $2.0\times10^{-1}$ | | Ebert et al. (2023) | ? | 365 |
| atraton<br>$C_9H_{17}N_5O$<br>[1610-17-9]<br>PXWUKZGIHQRDHL-UHFFFAOYSA-N | $6.4\times10^3$<br>$1.1\times10^4$<br>$2.2\times10^3$ | | Hilal et al. (2008)<br>Abraham et al. (2007)<br>MacBean (2012a) | Q<br>Q<br>? | |
| 4-hydroxy-2,2,6,6-tetramethyl-1-piperidinyloxy<br>$C_9H_{18}NO_2$<br>[2226-96-2]<br>UZFMOKQJFYMBGY-UHFFFAOYSA-N | $3.3\times10^9$ | | HSDB (2015) | Q | 99 |
| 8-hydroxyquinoline<br>$C_9H_7NO$<br>[148-24-3]<br>MCJGNVYPOGVAJF-UHFFFAOYSA-N | $1.7\times10^1$<br>$1.7\times10^1$<br>$8.9\times10^1$<br>$5.5\times10^2$ | | Duchowicz et al. (2020)<br>HSDB (2015)<br>Duchowicz et al. (2020)<br>Maniere et al. (2011) | V<br>V<br>Q<br>? | <br><br><br>241, 165 |
| carbendazim<br>$C_9H_9N_3O_2$<br>[10605-21-7]<br>TWFZGCMQGLPBSX-UHFFFAOYSA-N | $4.7\times10^5$<br>$6.5\times10^5$ | | HSDB (2015)<br>Mackay et al. (2006d) | V<br>V | |
| metamitron<br>$C_{10}H_{10}N_4O$<br>[41394-05-2]<br>VHCNQEUWZYOAEV-UHFFFAOYSA-N | $1.0\times10^6$<br>$2.2\times10^6$<br>$4.5\times10^3$<br>$2.8\times10^7$<br>$1.6\times10^7$<br>$1.1\times10^7$ | | Barcelo and Hennion (1997)<br>Delgado and Alderete (2003)<br>Goodarzi et al. (2010)<br>Delgado and Alderete (2003)<br>Delgado and Alderete (2003)<br>Maniere et al. (2011) | X<br>C<br>Q<br>Q<br>Q<br>? | 567<br><br>568<br><br><br>241, 165 |





Table A4.5: Heterocycles with oxygen and nitrogen (C, H, O, N) (…continued)

| Substance Formula (Trivial Name) [CAS Registry Number] InChIKey | $H_s^{cp}$ (at $T^{\ominus}$) $\left[\dfrac{\text{mol}}{\text{m}^3\,\text{Pa}}\right]$ | $\dfrac{\text{d}\ln H_s^{cp}}{\text{d}(1/T)}$ [K] | Reference | Type | Note |
|---|---|---|---|---|---|
| pymetrozin $C_{10}H_{11}N_5O$ [123312-89-0] QHMTXANCGGJZRX-UHFFFAOYSA-N | $3.3\times10^5$ | | HSDB (2015) | V | |
| 3-oxo-N-phenylbutanamide $C_{10}H_{11}NO_2$ (acetoacetanilide) [102-01-2] DYRDKSSFIWVSNM-UHFFFAOYSA-N | $2.3\times10^6$ | | HSDB (2015) | Q | 99 |
| 2,3'-didehydro-3'-deoxythymidine (stavudine) $C_{10}H_{12}N_2O_4$ (stavudine) [3056-17-5] XNKLLVCARDGLGL-JGVFFNPUSA-N | $4.3\times10^9$ | | HSDB (2015) | Q | 99 |
| cotinine $C_{10}H_{12}N_2O$ [486-56-6] UIKROCXWUNQSPJ-UHFFFAOYSA-N | $3.0\times10^6$ | | HSDB (2015) | Q | 99 |
| 4-(N-nitroso-N-methylamino)-1-(3-pyridyl)-1-butanone $C_{10}H_{13}N_3O_2$ [64091-91-4] FLAQQSHRLBFIEZ-UHFFFAOYSA-N | $1.2\times10^8$ | | HSDB (2015) | Q | 99 |
| 9-(4-hydroxy-3-hydroxymethylbut-1-yl)guanine $C_{10}H_{15}N_5O_3$ (penciclovir) [39809-25-1] JNTOCHDNEULJHD-UHFFFAOYSA-N | $1.0\times10^{26}$ | | HSDB (2015) | Q | 99 |
| anatoxin A $C_{10}H_{15}NO$ [64285-06-9] SGNXVBOIDPPRJJ-UHFFFAOYSA-N | $1.5\times10^3$ | | HSDB (2015) | Q | 99 |
| dimetilan $C_{10}H_{16}N_4O_3$ [644-64-4] RDBIYWSVMRVKSG-UHFFFAOYSA-N | $2.4\times10^5$ | | HSDB (2015) | Q | 99 |
| isolan $C_{10}H_{17}N_3O_2$ [119-38-0] RNNBHZYEKNHLKT-UHFFFAOYSA-N | $4.9\times10^3$ | | HSDB (2015) | Q | 99 |



Table A4.5: Heterocycles with oxygen and nitrogen (C, H, O, N) (. . . continued)

| Substance<br>Formula<br>(Trivial Name)<br>[CAS Registry Number]<br>InChIKey | $H_s^{cp}$<br>(at $T^\ominus$)<br>$\left[\dfrac{\text{mol}}{\text{m}^3\,\text{Pa}}\right]$ | $\dfrac{\text{d}\ln H_s^{cp}}{\text{d}(1/T)}$<br><br>[K] | Reference | Type | Note |
|---|---|---|---|---|---|
| amicarbazone<br>$C_{10}H_{19}N_5O_2$<br>[129909-90-6]<br>ORFPWVRKFLOQHK-UHFFFAOYSA-N | $1.5\times10^7$ | | MacBean (2012b) | X | 350 |
| prometone<br>$C_{10}H_{19}N_5O$<br>(prometon)<br>[1610-18-0]<br>ISEUFVQQFVOBCY-UHFFFAOYSA-N | $1.1\times10^4$<br>$1.1\times10^4$<br>$1.1\times10^4$<br>$1.1\times10^2$<br>$6.9\times10^1$<br>$2.7\times10^3$<br>$5.1\times10^3$ | | HSDB (2015)<br>Mackay et al. (2006d)<br>Suntio et al. (1988)<br>Barcelo and Hennion (1997)<br>Goodarzi et al. (2010)<br>Hilal et al. (2008)<br>Abraham et al. (2007) | V<br>V<br>V<br>X<br>Q<br>Q<br>Q | <br><br>12<br>567<br>568 |
| secbumeton<br>$C_{10}H_{19}N_5O$<br>[26259-45-0]<br>ZJMZZNVGNSWOOM-UHFFFAOYSA-N | $2.8\times10^3$<br>$2.9\times10^3$<br>$5.0\times10^3$<br>$7.2\times10^3$<br>$2.7\times10^3$ | | Mackay et al. (2006d)<br>Suntio et al. (1988)<br>Hilal et al. (2008)<br>Abraham et al. (2007)<br>MacBean (2012a) | V<br>V<br>Q<br>Q<br>? | <br>12 |
| terbumeton<br>$C_{10}H_{19}N_5O$<br>[33693-04-8]<br>BCQMBFHBDZVHKU-UHFFFAOYSA-N | $2.1\times10^3$<br>$2.4\times10^3$<br>$1.6\times10^3$ | | Mackay et al. (2006d)<br>Hilal et al. (2008)<br>Abraham et al. (2007) | V<br>Q<br>Q | |
| kinetin<br>$C_{10}H_9N_5O$<br>[525-79-1]<br>QANMHLXAZMSUEX-UHFFFAOYSA-N | $8.2\times10^8$ | | HSDB (2015) | Q | 99 |
| isouron<br>$C_{10}H_{17}N_3O_2$<br>[55861-78-4]<br>JLLJHQLUZAKJFH-UHFFFAOYSA-N | $7.3\times10^4$<br>$3.3\times10^1$ | | Duchowicz et al. (2020)<br>Duchowicz et al. (2020) | V<br>Q | 186 |
| fuberidazole<br>$C_{11}H_8N_2O$<br>[3878-19-1]<br>UYJUZNLFJAWNEZ-UHFFFAOYSA-N | $1.9\times10^5$ | | Ebert et al. (2023) | ? | 318 |
| carbadox<br>$C_{11}H_{10}N_4O_4$<br>[6804-07-5]<br>OVGGLBAWFMIPPY-WUXMJOGZSA-N | $2.2\times10^{17}$ | | HSDB (2015) | Q | 99 |
| bendiocarb<br>$C_{11}H_{13}NO_4$<br>[22781-23-3]<br>XEGGRYVFLWGFHI-UHFFFAOYSA-N | $2.5\times10^2$<br>$2.5\times10^2$<br>$2.7\times10^2$<br>$2.0\times10^2$ | | Duchowicz et al. (2020)<br>HSDB (2015)<br>Mackay et al. (2006d)<br>Duchowicz et al. (2020) | V<br>V<br>V<br>Q | 186 |



Table A4.5: Heterocycles with oxygen and nitrogen (C, H, O, N) (. . . continued)

| Substance<br>Formula<br>(Trivial Name)<br>[CAS Registry Number]<br>InChIKey | $H_s^{cp}$<br>(at $T^\ominus$)<br>$\left[\dfrac{\text{mol}}{\text{m}^3\,\text{Pa}}\right]$ | $\dfrac{\text{d}\ln H_s^{cp}}{\text{d}(1/T)}$<br><br>[K] | Reference | Type | Note |
|---|---|---|---|---|---|
| 2,3,5-trimethylphenol,<br>methylcarbamate<br>$C_{11}H_{15}NO_2$<br>(2,3,5-trimethacarb)<br>[2655-15-4]<br>NYOKZHDTNBDPOB-UHFFFAOYSA-N | $4.5\times10^1$ | | HSDB (2015) | V | |
| butalbital<br>$C_{11}H_{16}N_2O_3$<br>[77-26-9]<br>UZVHFVZFNXBMQJ-UHFFFAOYSA-N | $1.6\times10^7$ | | HSDB (2015) | Q | 99 |
| dexrazoxane<br>$C_{11}H_{16}N_4O_4$<br>[24584-09-6]<br>BMKDZUISNHGIBY-ZETCQYMHSA-N | $4.7\times10^{13}$ | | HSDB (2015) | Q | 99 |
| pentobarbital<br>$C_{11}H_{18}N_2O_3$<br>[76-74-4]<br>WEXRUCMBJFQVBZ-UHFFFAOYSA-N | $1.2\times10^7$ | | HSDB (2015) | Q | 99 |
| pirimor<br>$C_{11}H_{18}N_4O_2$<br>(pirimicarb)<br>[23103-98-2]<br>YFGYUFNIOHWBOB-UHFFFAOYSA-N | $1.2\times10^4$<br>$3.1\times10^3$<br>$5.0\times10^3$<br>$5.9\times10^3$<br>$3.1\times10^3$<br>$3.4\times10^4$<br><br>$3.0\times10^4$<br><br>$3.0\times10^4$<br><br>$2.8\times10^4$ | | HSDB (2015)<br>Mackay et al. (2006d)<br>Siebers and Mattusch (1996)<br>Siebers et al. (1994)<br>Suntio et al. (1988)<br>Maniere et al. (2011)<br><br>Maniere et al. (2011)<br><br>Maniere et al. (2011)<br><br>Maniere et al. (2011) | V<br>V<br>V<br>V<br>V<br>?<br><br>?<br><br>?<br><br>? | <br><br>12<br><br>12<br>12, 574,<br>165<br>12, 575,<br>165<br>12, 576,<br>165<br>12, 165 |
| ethirimol<br>$C_{11}H_{19}N_3O$<br>[23947-60-6]<br>BBXXLROWFHWFQY-UHFFFAOYSA-N | $3.6\times10^3$ | | Mackay et al. (2006d) | V | |
| pyroquilon<br>$C_{11}H_{11}NO$<br>[57369-32-1]<br>XRJLAOUDSILTFT-UHFFFAOYSA-N | $5.1\times10^3$ | | Ebert et al. (2023) | ? | 318 |
| fenfuram<br>$C_{12}H_{11}NO_2$<br>[24691-80-3]<br>JFSPBVWPKOEZCB-UHFFFAOYSA-N | $2.5\times10^4$ | | Mackay et al. (2006d) | V | |



Table A4.5: Heterocycles with oxygen and nitrogen (C, H, O, N) (...continued)

| Substance<br>Formula<br>(Trivial Name)<br>[CAS Registry Number]<br>InChIKey | $H_s^{cp}$<br>(at $T^{\ominus}$)<br>$\left[\dfrac{\text{mol}}{\text{m}^3\,\text{Pa}}\right]$ | $\dfrac{\text{d}\ln H_s^{cp}}{\text{d}(1/T)}$<br><br>[K] | Reference | Type | Note |
|---|---|---|---|---|---|
| oxabetrinil<br>$C_{12}H_{12}N_2O_3$<br>[74782-23-3]<br>WFVUIONFJOAYPK-UHFFFAOYSA-N | $8.6\times10^1$ | | Ebert et al. (2023) | ? | 318 |
| phenobarbital<br>$C_{12}H_{12}N_2O_3$<br>[50-06-6]<br>DDBREPKUVSBGFI-UHFFFAOYSA-N | $5.8\times10^8$ | | HSDB (2015) | Q | 99 |
| triaziquone<br>$C_{12}H_{13}N_3O_2$<br>[68-76-8]<br>PXSOHRWMIRDKMP-UHFFFAOYSA-N | $1.1\times10^{10}$ | | HSDB (2015) | Q | 99 |
| triallyl cyanurate<br>$C_{12}H_{15}N_3O_3$<br>[101-37-1]<br>BJELTSYBAHKXRW-UHFFFAOYSA-N | $2.3\times10^1$<br>$1.8\times10^3$<br>$1.9\times10^2$<br>$4.1\times10^4$ | | Zhang et al. (2010)<br>Zhang et al. (2010)<br>Zhang et al. (2010)<br>Zhang et al. (2010) | Q<br>Q<br>Q<br>Q | 287, 288<br>287, 289<br>287, 290<br>287, 291 |
| entecavir<br>$C_{12}H_{15}N_5O_3$<br>[142217-69-4]<br>QDGZDCVAUDNJFG-FXQIFTODSA-N | $6.2\times10^{15}$ | | HSDB (2015) | Q | 99 |
| metaxalone<br>$C_{12}H_{15}NO_3$<br>[1665-48-1]<br>IMWZZHHPURKASS-UHFFFAOYSA-N | $3.7\times10^4$ | | HSDB (2015) | Q | 99 |
| phendimetrazine<br>$C_{12}H_{17}NO$<br>[634-03-7]<br>MFOCDFTXLCYLKU-UHFFFAOYSA-N | $3.7\times10^2$ | | HSDB (2015) | Q | 99 |
| hexazinone<br>$C_{12}H_{20}N_4O_2$<br>[51235-04-2]<br>CAWXEEYDBZRFPE-UHFFFAOYSA-N | $>9.9\times10^1$<br>$4.4\times10^6$ | | Mabury and Crosby (1996)<br>HSDB (2015) | M<br>V | |
| picaridin<br>$C_{12}H_{23}NO_3$<br>[119515-38-7]<br>QLHULAHOXSSASE-UHFFFAOYSA-N | $3.3\times10^5$ | | HSDB (2015) | Q | 99 |
| lenacil<br>$C_{13}H_{18}N_2O_2$<br>[2164-08-1]<br>ZTMKADLOSYKWCA-UHFFFAOYSA-N | $1.3\times10^5$<br>$3.8\times10^4$<br>$7.7\times10^6$ | | Duchowicz et al. (2020)<br>Duchowicz et al. (2020)<br>Maniere et al. (2011) | V<br>Q<br>? | 186<br><br>241, 165 |





Table A4.5: Heterocycles with oxygen and nitrogen (C, H, O, N) (. . . continued)

| Substance Formula (Trivial Name) [CAS Registry Number] InChIKey | $H_s^{cp}$ (at $T^{\ominus}$) $\left[\dfrac{\text{mol}}{\text{m}^3\,\text{Pa}}\right]$ | $\dfrac{\text{d}\ln H_s^{cp}}{\text{d}(1/T)}$ [K] | Reference | Type | Note |
|---|---|---|---|---|---|
| pyrinuron $C_{13}H_{12}N_4O_3$ (pyriminil) [53558-25-1] CLKZWXHKFXZIMA-UHFFFAOYSA-N | $5.4\times10^{10}$ | | HSDB (2015) | Q | 99 |
| melatonin $C_{13}H_{16}N_2O_2$ [73-31-4] DRLFMBDRBRZALE-UHFFFAOYSA-N | $3.8\times10^{8}$ | | HSDB (2015) | Q | 447 |
| dibenz[$b,f$]][1,4]oxazepine $C_{13}H_9NO$ [257-07-8] NPUACKRELIJTFM-UHFFFAOYSA-N | $2.4\times10^{-3}$ | | HSDB (2015) | Q | 99 |
| oxadixyl $C_{14}H_{18}N_2O_4$ [77732-09-3] UWVQIROCRJWDKL-UHFFFAOYSA-N | $2.0\times10^{6}$ | | Ebert et al. (2023) | ? | 318 |
| benomyl $C_{14}H_{18}N_4O_3$ [17804-35-2] RIOXQFHNBCKOKP-UHFFFAOYSA-N | $5.2\times10^{5}$ $3.1\times10^{5}$ $1.8\times10^{5}$ $2.0\times10^{6}$ | | Mackay et al. (2006d) Keshavarz et al. (2022) Duchowicz et al. (2020) Duchowicz et al. (2020) | V Q Q ? | 185, 21 |
| trimethoprim $C_{14}H_{18}N_4O_3$ [738-70-5] IEDVJHCEMCRBQM-UHFFFAOYSA-N | $4.1\times10^{8}$ | | HSDB (2015) | Q | 99 |
| ethoxyquin $C_{14}H_{19}NO$ [91-53-2] DECIPOUIJURFOJ-UHFFFAOYSA-N | 8.0 | | Ebert et al. (2023) | ? | 318 |
| famciclovir $C_{14}H_{19}N_5O_4$ [104227-87-4] GGXKWVWZWMLJEH-UHFFFAOYSA-N | $1.0\times10^{8}$ | | HSDB (2015) | Q | 99 |
| furmecyclox $C_{14}H_{21}NO_3$ [60568-05-0] QTDRLOKFLJJHTG-UHFFFAOYSA-N | $1.4\times10^{2}$ | | MacBean (2012a) | ? | |
| oxcarbazepine $C_{15}H_{12}N_2O_2$ [28721-07-5] CTRLABGOLIVAIY-UHFFFAOYSA-N | $1.4\times10^{7}$ | | HSDB (2015) | Q | 99 |





Table A4.5: Heterocycles with oxygen and nitrogen (C, H, O, N) (. . . continued)

| Substance<br>Formula<br>(Trivial Name)<br>[CAS Registry Number]<br>InChIKey | $H_s^{cp}$<br>(at $T^{\ominus}$)<br>$\left[\dfrac{\mathrm{mol}}{\mathrm{m^3\,Pa}}\right]$ | $\dfrac{\mathrm{d}\ln H_s^{cp}}{\mathrm{d}(1/T)}$<br><br>[K] | Reference | Type | Note |
|---|---|---|---|---|---|
| phenytoin<br>$C_{15}H_{12}N_2O_2$<br>[57-41-0]<br>CXOFVDLJLONNDW-UHFFFAOYSA-N | $9.7{\times}10^5$ | | HSDB (2015) | Q | 99 |
| carbamazepine<br>$C_{15}H_{12}N_2O$<br>[298-46-4]<br>FFGPTBGBLSHEPO-UHFFFAOYSA-N | $9.0{\times}10^4$ | | HSDB (2015) | Q | 99 |
| propylthiouracil<br>$C_{15}H_{12}N_2O$<br>[51-52-5]<br>KNAHARQHSZJURB-UHFFFAOYSA-N | $9.0{\times}10^3$ | | HSDB (2015) | Q | 99 |
| ancymidol<br>$C_{15}H_{16}N_2O_2$<br>[12771-68-5]<br>HUTDUHSNJYTCAR-UHFFFAOYSA-N | $9.1{\times}10^5$<br>$1.5{\times}10^6$<br>$4.7{\times}10^6$<br>$6.3{\times}10^5$ | | Keshavarz et al. (2022)<br>Duchowicz et al. (2020)<br>Hilal et al. (2008)<br>Duchowicz et al. (2020) | Q<br>Q<br>Q<br>? | <br><br><br>185, 21 |
| imazethapyr<br>$C_{15}H_{19}N_3O_3$<br>[81335-77-5]<br>XVOKUMIPKHGGTN-UHFFFAOYSA-N | $9.9{\times}10^{10}$ | | HSDB (2015) | Q | 99 |
| imazamox<br>$C_{15}H_{19}N_3O_4$<br>[114311-32-9]<br>NUPJIGQFXCQJBK-UHFFFAOYSA-N | $1.1{\times}10^{13}$<br>$2.0{\times}10^{11}$ | | HSDB (2015)<br>Maniere et al. (2011) | Q<br>? | 99<br>241, 165 |
| cycloheximide<br>$C_{15}H_{23}NO_4$<br>[66-81-9]<br>YPHMISFOHDHNIV-FSZOTQKASA-N | $2.8{\times}10^9$ | | HSDB (2015) | Q | 99 |
| oxymatrine<br>$C_{15}H_{24}N_2O_2$<br>[16837-52-8]<br>XVPBINOPNYFXID-LHDUFFHYSA-N | $9.9{\times}10^{12}$ | | HSDB (2015) | Q | 99 |
| triapenthenol<br>$C_{15}H_{25}N_3O$<br>[76608-88-3]<br>CNFMJLVJDNGPHR-UKTHLTGXSA-N | $4.0{\times}10^4$ | | Ebert et al. (2023) | ? | 318 |
| mebendazole<br>$C_{16}H_{13}N_3O_3$<br>[31431-39-7]<br>OPXLLQIJSORQAM-UHFFFAOYSA-N | $1.8{\times}10^{10}$ | | HSDB (2015) | Q | 99 |



Table A4.5: Heterocycles with oxygen and nitrogen (C, H, O, N) (...continued)

| Substance Formula (Trivial Name) [CAS Registry Number] InChIKey | $H_s^{cp}$ (at $T^\ominus$) $\left[\dfrac{\text{mol}}{\text{m}^3\,\text{Pa}}\right]$ | $\dfrac{\text{d}\ln H_s^{cp}}{\text{d}(1/T)}$ [K] | Reference | Type | Note |
|---|---|---|---|---|---|
| imazamethabenz-methyl $C_{16}H_{20}N_2O_3$ [81405-85-8] FFCCBBNQPIMUJI-UHFFFAOYSA-N | $2.6\times10^6$ | | HSDB (2015) | V | |
| nifedipine $C_{17}H_{18}N_2O_6$ [21829-25-4] HYIMSNHJOBLJNT-UHFFFAOYSA-N | $1.4\times10^8$ | | HSDB (2015) | Q | 99 |
| oxymorphone $C_{17}H_{19}NO_4$ [76-41-5] UQCNKQCJZOAFTQ-UHFFFAOYSA-N | $2.4\times10^{13}$ | | HSDB (2015) | Q | 99 |
| desomorphine $C_{17}H_{21}NO_2$ [427-00-9] LNNWVNGFPYWNQE-UHFFFAOYSA-N | $2.4\times10^6$ | | HSDB (2015) | Q | 99 |
| cocaine $C_{17}H_{21}NO_4$ [50-36-2] ZPUCINDJVBIVPJ-PFSRBDOWSA-N | $2.3\times10^5$ $2.3\times10^5$ $7.9\times10^4$ | | Duchowicz et al. (2020) HSDB (2015) Duchowicz et al. (2020) | V V Q | 186 |
| N-(2-ethylhexyl)-5-norbornene-2,3-dicarboximide $C_{17}H_{25}NO_2$ [113-48-4] WLLGXSLBOPFWQV-UHFFFAOYSA-N | $3.5\times10^1$ | | HSDB (2015) | Q | 99 |
| imazaquin $C_{17}H_{17}N_3O_3$ [81335-37-7] CABMTIJINOIHOD-UHFFFAOYSA-N | $2.7\times10^{11}$ | | Maniere et al. (2011) | ? | 12, 165 |
| (E)-pyriminobac-methyl $C_{17}H_{19}N_3O_6$ [147411-69-6] USSIUIGPBLPCDF-KEBDBYFISA-N | $7.3\times10^2$ | | Ebert et al. (2023) | ? | 318 |
| (Z)-pyriminobac-methyl $C_{17}H_{19}N_3O_6$ [147411-70-9] USSIUIGPBLPCDF-JMIUGGIZSA-N | $1.9\times10^4$ | | Ebert et al. (2023) | ? | 318 |
| imiprothrin $C_{17}H_{22}N_2O_4$ [72963-72-5] VPRAQYXPZIFIOH-UHFFFAOYSA-N | $1.6\times10^5$ $6.9\times10^3$ | | Duchowicz et al. (2020) Duchowicz et al. (2020) | V Q | 186 |



**Rolf Sander: Compilation of Henry's law constants** 829

Table A4.5: Heterocycles with oxygen and nitrogen (C, H, O, N) (...continued)

| Substance<br>Formula<br>(Trivial Name)<br>[CAS Registry Number]<br><small>InChIKey</small> | $H_s^{cp}$<br>(at $T^{\ominus}$)<br>$\left[\dfrac{\text{mol}}{\text{m}^3\,\text{Pa}}\right]$ | $\dfrac{\text{d}\ln H_s^{cp}}{\text{d}(1/T)}$<br><br>[K] | Reference | Type | Note |
|---|---|---|---|---|---|
| iprovalicarb<br>$C_{18}H_{28}N_2O_3$<br>[140923-17-7]<br><small>NWUWYYSKZYIQAE-LBAUFKAWSA-N</small> | $7.1 \times 10^5$ | | Maniere et al. (2011) | ? | 12, 165 |
| quinophthalone<br>$C_{18}H_{11}NO_2$<br>[8003-22-3]<br><small>IZMJMCDDWKSTTK-UHFFFAOYSA-N</small> | $1.6 \times 10^8$ | | HSDB (2015) | Q | 99 |
| pefurazoate<br>$C_{18}H_{23}N_3O_4$<br>[101903-30-4]<br><small>WBTYBAGIHOISOQ-UHFFFAOYSA-N</small> | $1.8 \times 10^3$ | | Ebert et al. (2023) | ? | 318 |
| tetramethrin<br>$C_{19}H_{25}NO_4$<br>[7696-12-0]<br><small>CXBMCYHAMVGWJQ-UHFFFAOYSA-N</small> | $5.8$ | | HSDB (2015) | V | |
| isoxaben<br>$C_{19}H_{25}NO_4$<br>[82558-50-7]<br><small>PMHURSZHKKJGBM-UHFFFAOYSA-N</small> | $7.8 \times 10^3$<br>$7.8 \times 10^3$<br>$5.3 \times 10^3$<br>$5.1 \times 10^3$ | | Duchowicz et al. (2020)<br>MacBean (2012b)<br>Duchowicz et al. (2020)<br>Maniere et al. (2011) | V<br>X<br>Q<br>? | 186<br>350<br><br>12, 165 |
| alfuzosin<br>$C_{19}H_{27}N_5O_4$<br>[81403-80-7]<br><small>WNMJYKCGWZFFKR-UHFFFAOYSA-N</small> | $1.0 \times 10^{14}$ | | HSDB (2015) | Q | 99 |
| 2,6-dimethyl-4-tridecylmorpholine<br>$C_{19}H_{39}NO$<br>(tridemorph)<br>[24602-86-6]<br><small>YTOPFCCWCSOHFV-UHFFFAOYSA-N</small> | $5.8 \times 10^{-2}$ | | Ebert et al. (2023) | ? | 318 |
| pyriproxyfen<br>$C_{20}H_{19}NO_3$<br>[95737-68-1]<br><small>NHDHVHZZCFYRSB-UHFFFAOYSA-N</small> | $1.6 \times 10^4$<br>$>1.4 \times 10^1$ | | HSDB (2015)<br>Maniere et al. (2011) | Q<br>? | 99<br>72, 165 |
| papaverine<br>$C_{20}H_{21}NO_4$<br>[58-74-2]<br><small>XQYZDYMELSJDRZ-UHFFFAOYSA-N</small> | $1.3 \times 10^7$ | | HSDB (2015) | Q | 99 |
| fenazaquin<br>$C_{20}H_{22}N_2O$<br>[120928-09-8]<br><small>DMYHGDXADUDKCQ-UHFFFAOYSA-N</small> | $2.1 \times 10^2$<br>$9.9 \times 10^1$<br>$4.7 \times 10^1$<br>$1.8 \times 10^1$ | | Duchowicz et al. (2020)<br>HSDB (2015)<br>Duchowicz et al. (2020)<br>Maniere et al. (2011) | V<br>V<br>Q<br>? | 186<br><br><br>241, 165 |



Table A4.5: Heterocycles with oxygen and nitrogen (C, H, O, N) (...continued)

| Substance<br>Formula<br>(Trivial Name)<br>[CAS Registry Number]<br>InChIKey | $H_s^{cp}$<br>(at $T^{\ominus}$)<br>$\left[\dfrac{\text{mol}}{\text{m}^3\,\text{Pa}}\right]$ | $\dfrac{\text{d}\ln H_s^{cp}}{\text{d}(1/T)}$<br><br>[K] | Reference | Type | Note |
|---|---|---|---|---|---|
| bitertanol<br>$C_{20}H_{23}N_3O_2$<br>[55179-31-2]<br>VGPIBGGRCVEHQZ-UHFFFAOYSA-N | $1.2\times10^4$ | | Mackay et al. (2006d) | V | |
| bitertanol diastereoisomer a<br>$C_{20}H_{23}N_3O_2$<br>[70585-36-3]<br>VGPIBGGRCVEHQZ-OALUTQOASA-N | $3.1\times10^6$ | | Mackay et al. (2006d) | V | |
| bitertanol diastereoisomer b<br>$C_{20}H_{23}N_3O_2$<br>[70585-38-5]<br>VGPIBGGRCVEHQZ-RBUKOAKNSA-N | $1.5\times10^6$ | | Mackay et al. (2006d) | V | |
| naltrexone<br>$C_{20}H_{23}NO_4$<br>[16590-41-3]<br>DQCKKXVULJGBQN-UHFFFAOYSA-N | $2.3\times10^{13}$ | | HSDB (2015) | Q | 99 |
| $D$-lysergic acid N,N-diethylamide<br>$C_{20}H_{25}N_3O$<br>(LSD)<br>[50-37-3]<br>VAYOSLLFUXYJDT-UHFFFAOYSA-N | $6.6\times10^{10}$ | | HSDB (2015) | Q | 99 |
| ibogaine<br>$C_{20}H_{26}N_2O$<br>[83-74-9]<br>HSIBGVUMFOSJPD-UHFFFAOYSA-N | $8.2\times10^5$ | | HSDB (2015) | Q | 99 |
| fenpropimorph<br>$C_{20}H_{33}NO$<br>[67564-91-4]<br>RYAUSSKQMZRMAI-ALOPSCKCSA-N | 6.2<br>$6.2\times10^{-2}$<br>$7.9\times10^{-2}$<br>$3.6\times10^{-1}$ | | Mackay et al. (2006d)<br>Barcelo and Hennion (1997)<br>Goodarzi et al. (2010)<br>Maniere et al. (2011) | V<br>X<br>Q<br>? | <br>567<br>568<br>241, 165 |
| strychnine<br>$C_{21}H_{22}N_2O_2$<br>[57-24-9]<br>QMGVPVSNSZLJIA-UHFFFAOYSA-N | $1.6\times10^8$ | | HSDB (2015) | Q | 99 |
| nalmefene<br>$C_{21}H_{25}NO_3$<br>[55096-26-9]<br>WJBLNOPPDWQMCH-UHFFFAOYSA-N | $5.5\times10^{10}$ | | HSDB (2015) | Q | 99 |
| benztropine<br>$C_{21}H_{25}NO$<br>[86-13-5]<br>GIJXKZJWITVLHI-UHFFFAOYSA-N | $4.5\times10^3$ | | HSDB (2015) | Q | 99 |



Table A4.5: Heterocycles with oxygen and nitrogen (C, H, O, N) (...continued)

| Substance Formula (Trivial Name) [CAS Registry Number] InChIKey | $H_s^{cp}$ (at $T^\ominus$) $\left[\dfrac{\text{mol}}{\text{m}^3\,\text{Pa}}\right]$ | $\dfrac{\text{d}\ln H_s^{cp}}{\text{d}(1/T)}$ [K] | Reference | Type | Note |
|---|---|---|---|---|---|
| stanozolol $C_{21}H_{32}N_2O$ [10418-03-8] LKAJKIOFIWVMDJ-KIWJEFSTSA-N | $9.0\times10^2$ | | HSDB (2015) | Q | 99 |
| spirotetramat | $1.4\times10^7$ | | Maniere et al. (2011) | ? | 241, 493, 165 |
| $C_{21}H_{27}NO_5$ | $1.6\times10^7$ | | Maniere et al. (2011) | ? | 241, 577, 165 |
| [203313-25-1] CLSVJBIHYWPGQY-GGYDESQDSA-N | $9.2\times10^6$ | | Maniere et al. (2011) | ? | 241, 573, 165 |
| diacetylmorphine $C_{21}H_{23}NO_5$ (heroin) [561-27-3] GVGLGOZIDCSQPN-PVHGPHFFSA-N | $1.6\times10^7$ $2.7\times10^5$ | | Duchowicz et al. (2020) Duchowicz et al. (2020) | V Q | 186 |
| pyrametostrobin $C_{21}H_{23}N_3O_4$ [915410-70-7] DWTVBEZBWMDXIY-UHFFFAOYSA-N | $2.2\times10^4$ | | Ebert et al. (2023) | ? | 318 |
| azoxystrobin $C_{22}H_{17}N_3O_5$ [131860-33-8] WFDXOXNFNRHQEC-GHRIWEEISA-N | $1.4\times10^8$ $1.4\times10^8$ | | HSDB (2015) Maniere et al. (2011) | V ? | 241, 165 |
| famoxadone $C_{22}H_{18}N_2O_4$ [131807-57-3] PCCSBWNGDMYFCW-UHFFFAOYSA-N | $2.1\times10^2$ $2.2\times10^2$ | | HSDB (2015) Maniere et al. (2011) | V ? | 241, 165 |
| tadalafil $C_{22}H_{19}N_3O_4$ [171596-29-5] WOXKDUGGOYFFRN-IIBYNOLFSA-N | $2.0\times10^{12}$ | | HSDB (2015) | Q | 99 |
| bisacodyl $C_{22}H_{19}NO_4$ [603-50-9] KHOITXIGCFIULA-UHFFFAOYSA-N | $1.4\times10^6$ | | HSDB (2015) | Q | 99 |
| fentanyl $C_{22}H_{28}N_2O$ [437-38-7] PJMPHNIQZUBGLI-UHFFFAOYSA-N | $1.1\times10^6$ | | HSDB (2015) | Q | 99 |



Table A4.5: Heterocycles with oxygen and nitrogen (C, H, O, N) (...continued)

| Substance Formula (Trivial Name) [CAS Registry Number] InChIKey | $H_s^{cp}$ (at $T^\ominus$) $\left[\dfrac{\text{mol}}{\text{m}^3\,\text{Pa}}\right]$ | $\dfrac{\text{d}\ln H_s^{cp}}{\text{d}(1/T)}$ [K] | Reference | Type | Note |
|---|---|---|---|---|---|
| 4-(triphenylmethyl)morpholine C$_{23}$H$_{23}$NO (trifenmorph) [1420-06-0] ZJMLMBICUVVJDX-UHFFFAOYSA-N | $7.6 \times 10^4$ 3.2 | | HSDB (2015) MacBean (2012a) | Q ? | 99 |
| brucine C$_{23}$H$_{26}$N$_2$O$_4$ [357-57-3] RRKTZKIUPZVBMF-UHFFFAOYSA-N | $4.7 \times 10^{10}$ | | HSDB (2015) | Q | 99 |
| mycophenolate mofetil C$_{23}$H$_{31}$NO$_7$ [128794-94-5] RTGDFNSFWBGLEC-SYZQJQIISA-N | $1.8 \times 10^9$ | | HSDB (2015) | Q | 99 |
| pinoxaden C$_{23}$H$_{32}$N$_2$O$_4$ [243973-20-8] MGOHCFMYLBAPRN-UHFFFAOYSA-N | $1.1 \times 10^6$ $1.1 \times 10^6$ | | HSDB (2015) Maniere et al. (2011) | V ? | 165 |
| fenpyroximate C$_{24}$H$_{27}$N$_3$O$_4$ [134098-61-6] YYJNOYZRYGDPNH-MFKUBSTISA-N | 4.6 7.6 $5.8 \times 10^4$ | | Duchowicz et al. (2020) MacBean (2012b) Duchowicz et al. (2020) | V X Q | 186 350 |
| valsartan C$_{24}$H$_{29}$N$_5$O$_3$ [137862-53-4] ACWBQPMHZXGDFX-QFIPXVFZSA-N | $3.2 \times 10^{12}$ | | HSDB (2015) | Q | 99 |
| donepezil C$_{24}$H$_{29}$NO$_3$ [120014-06-4] ADEBPBSSDYVVLD-UHFFFAOYSA-N | $8.2 \times 10^6$ | | HSDB (2015) | Q | 99 |
| chromafenozide C$_{24}$H$_{30}$N$_2$O$_3$ [143807-66-3] HPNSNYBUADCFDR-UHFFFAOYSA-N | $5.1 \times 10^5$ $6.2 \times 10^5$ $5.6 \times 10^5$ | | Maniere et al. (2011) Maniere et al. (2011) Maniere et al. (2011) | ? ? ? | 12, 493, 165 12, 577, 165 12, 573, 165 |
| cyenopyrafen C$_{24}$H$_{31}$N$_3$O$_2$ [560121-52-0] APJLTUBHYCOZJI-VZCXRCSSSA-N | $1.6 \times 10^3$ | | Ebert et al. (2023) | ? | 318 |





Table A4.5: Heterocycles with oxygen and nitrogen (C, H, O, N) (...continued)

| Substance Formula (Trivial Name) [CAS Registry Number] InChIKey | $H_s^{cp}$ (at $T^\ominus$) $\left[\dfrac{\text{mol}}{\text{m}^3\,\text{Pa}}\right]$ | $\dfrac{\text{d}\ln H_s^{cp}}{\text{d}(1/T)}$ [K] | Reference | Type | Note |
|---|---|---|---|---|---|
| 2-[4-[4-(2-benzoxazolyl)styryl]phenyl]-5-methylbenzoxazole | $7.5\times10^{8}$ | | Zhang et al. (2010) | Q | 287, 288 |
| $C_{29}H_{20}N_2O_2$ | $6.2\times10^{6}$ | | Zhang et al. (2010) | Q | 287, 289 |
| [5242-49-9] | $1.2\times10^{5}$ | | Zhang et al. (2010) | Q | 287, 290 |
| SOTPOQQKAUOHRO-BQYQJAHWSA-N | $9.5\times10^{7}$ | | Zhang et al. (2010) | Q | 287, 291 |
| 2-(2H-benzotriazol-2-yl)-4,6-bis(1-methyl-1-phenylethyl)phenol | $7.2\times10^{9}$ | | Zhang et al. (2010) | Q | 287, 288 |
| $C_{30}H_{29}N_3O$ | $5.8\times10^{5}$ | | Zhang et al. (2010) | Q | 287, 289 |
| [70321-86-7] | $1.4\times10^{7}$ | | Zhang et al. (2010) | Q | 287, 290 |
| OLFNXLXEGXRUOI-UHFFFAOYSA-N | $8.8\times10^{6}$ | | Zhang et al. (2010) | Q | 287, 291 |
| fenpicoxamid $C_{31}H_{38}N_2O_{11}$ [517875-34-2] QGTOTYJSCYHYFK-RBODFLQRSA-N | $4.2\times10^{2}$ | | Maniere et al. (2011) | ? | 165 |
| norbormide $C_{33}H_{25}N_3O_3$ [991-42-4] DNTHHIVFNQZZRD-UHFFFAOYSA-N | $3.7\times10^{17}$ | | HSDB (2015) | Q | 99 |
| reserpine $C_{33}H_{40}N_2O_9$ [50-55-5] QEVHRUUCFGRFIF-UHFFFAOYSA-N | $1.8\times10^{17}$ | | HSDB (2015) | Q | 99 |
| telmisartan $C_{33}H_{30}N_4O_2$ [144701-48-4] RMMXLENWKUUMAY-UHFFFAOYSA-N | $1.9\times10^{14}$ | | Abraham et al. (2019) | Q | |
| telaprevir $C_{36}H_{53}N_7O_6$ [402957-28-2] BBAWEDCPNXPBQM-GDEBMMAJSA-N | $1.3\times10^{25}$ | | HSDB (2015) | Q | 99 |
| lopinavir $C_{37}H_{48}N_4O_5$ [192725-17-0] KJHKTHWMRKYKJE-WRHCQWCJSA-N | $2.3\times10^{22}$ | | HSDB (2015) | Q | 99 |
| atazanavir $C_{38}H_{52}N_6O_7$ [198904-31-3] AXRYRYVKAWYZBR-GASGPIRDSA-N | $2.7\times10^{26}$ | | HSDB (2015) | Q | 99 |



Table A4.5: Heterocycles with oxygen and nitrogen (C, H, O, N) (. . . continued)

| Substance<br>Formula<br>(Trivial Name)<br>[CAS Registry Number]<br><small>InChIKey</small> | $H_s^{cp}$ (at $T^{\ominus}$) $\left[\dfrac{\text{mol}}{\text{m}^3\,\text{Pa}}\right]$ | $\dfrac{\mathrm{d}\ln H_s^{cp}}{\mathrm{d}(1/T)}$ [K] | Reference | Type | Note |
|---|---|---|---|---|---|
| tylosin<br>$C_{46}H_{77}NO_{17}$<br>[1401-69-0]<br><small>WBPYTXDJUQJLPQ-SNQVITFCSA-N</small> | $1.7\times10^{32}$ | | HSDB (2015) | Q | 99 |
| nystatin<br>$C_{47}H_{75}NO_{17}$<br>[1400-61-9]<br><small>VQOXZBDYSJBXMA-QFHUWGMOSA-N</small> | $4.9\times10^{4}$ | | HSDB (2015) | Q | 99 |
| 1,3,5-tris(3,5-di-*tert*-butyl-4-<br>hydroxybenzyl)-1,3,5-triazinane-<br>2,4,6-trione<br>$C_{48}H_{69}N_3O_6$<br>[27676-62-6]<br><small>VNQNXQYZMPJLQX-UHFFFAOYSA-N</small> | $6.1\times10^{20}$<br><br>$1.3\times10^{12}$<br>$3.4\times10^{10}$<br>$8.2\times10^{14}$ | | Zhang et al. (2010)<br><br>Zhang et al. (2010)<br>Zhang et al. (2010)<br>Zhang et al. (2010) | Q<br><br>Q<br>Q<br>Q | 287, 288<br><br>287, 289<br>287, 290<br>287, 291 |



### A4.6 Nitrates ($RONO_2$)

Table A4.6: Nitrates ($RONO_2$)

| Substance Formula (Trivial Name) [CAS Registry Number] InChIKey | $H_s^{cp}$ (at $T^{\ominus}$) $\left[\dfrac{mol}{m^3\,Pa}\right]$ | $\dfrac{d\ln H_s^{cp}}{d(1/T)}$ [K] | Reference | Type | Note |
|---|---|---|---|---|---|
| urea nitrate $CH_5N_3O_4$ [124-47-0] AYTGUZPQPXGYFS-UHFFFAOYSA-N | $5.8\times10^{11}$ | | HSDB (2015) | Q | 99 |
| methyl nitrate $CH_3ONO_2$ [598-58-3] LRMHVVPPGGOAJQ-UHFFFAOYSA-N | $2.0\times10^{-2}$ | 4700 | Burkholder et al. (2019) | L | |
| | $2.0\times10^{-2}$ | 4700 | Burkholder et al. (2015) | L | |
| | $2.0\times10^{-2}$ | 4700 | Sander et al. (2011) | L | |
| | $2.0\times10^{-2}$ | 4700 | Sander et al. (2006) | L | |
| | $2.0\times10^{-2}$ | 4700 | Kames and Schurath (1992) | M | |
| | $2.6\times10^{-2}$ | | Schwartz (1986) | C | 87 |
| | $1.6\times10^{-2}$ | | Wang et al. (2017) | Q | 80, 238 |
| | $1.3\times10^{-1}$ | | Wang et al. (2017) | Q | 80, 239 |
| | $1.0\times10^{-3}$ | | Wang et al. (2017) | Q | 80, 240 |
| | $1.2\times10^{-2}$ | | Raventos-Duran et al. (2010) | Q | 242, 243 |
| | $6.2\times10^{-2}$ | | Raventos-Duran et al. (2010) | Q | 244 |
| | $4.9\times10^{-2}$ | | Raventos-Duran et al. (2010) | Q | 245 |
| | $6.2\times10^{-2}$ | | Hilal et al. (2008) | Q | |
| | | 4900 | Kühne et al. (2005) | Q | |
| | | 4800 | Kühne et al. (2005) | ? | |
| ethyl nitrate $C_2H_5ONO_2$ [625-58-1] IDNUEBSJWINEMI-UHFFFAOYSA-N | $1.6\times10^{-2}$ | 5400 | Burkholder et al. (2019) | L | |
| | $1.6\times10^{-2}$ | 5400 | Burkholder et al. (2015) | L | |
| | $1.6\times10^{-2}$ | 5400 | Sander et al. (2011) | L | |
| | $1.6\times10^{-2}$ | 5400 | Sander et al. (2006) | L | |
| | $1.6\times10^{-2}$ | 5400 | Kames and Schurath (1992) | M | |
| | $1.2\times10^{-2}$ | | Wang et al. (2017) | Q | 80, 238 |
| | $6.5\times10^{-2}$ | | Wang et al. (2017) | Q | 80, 239 |
| | $9.1\times10^{-4}$ | | Wang et al. (2017) | Q | 80, 240 |
| | $3.3\times10^{-2}$ | | HSDB (2015) | Q | 99 |
| | $9.9\times10^{-3}$ | | Raventos-Duran et al. (2010) | Q | 242, 243 |
| | $3.9\times10^{-2}$ | | Raventos-Duran et al. (2010) | Q | 244 |
| | $3.9\times10^{-2}$ | | Raventos-Duran et al. (2010) | Q | 245 |
| | $3.9\times10^{-2}$ | | Hilal et al. (2008) | Q | |
| 1-propyl nitrate $C_3H_7ONO_2$ [627-13-4] JNTOKFNBDFMTIV-UHFFFAOYSA-N | $1.1\times10^{-2}$ | 5500 | Burkholder et al. (2019) | L | |
| | $1.1\times10^{-2}$ | 5500 | Burkholder et al. (2015) | L | |
| | $1.1\times10^{-2}$ | 5500 | Sander et al. (2011) | L | |
| | $1.1\times10^{-2}$ | 5500 | Sander et al. (2006) | L | |
| | $9.0\times10^{-3}$ | 5600 | Staudinger and Roberts (2001) | L | |
| | $7.4\times10^{-3}$ | 4600 | Hauff et al. (1998) | M | |
| | $1.1\times10^{-2}$ | 5500 | Kames and Schurath (1992) | M | |
| | $1.1\times10^{-2}$ | | Hauff et al. (1998) | V | |
| | $3.6\times10^{-3}$ | | Keshavarz et al. (2022) | Q | |
| | $6.9\times10^{-3}$ | | Duchowicz et al. (2020) | Q | 184 |
| | $1.0\times10^{-2}$ | | Wang et al. (2017) | Q | 80, 238 |





Table A4.6: Nitrates ($RONO_2$) (... continued)

| Substance Formula (Trivial Name) [CAS Registry Number] InChIKey | $H_s^{cp}$ (at $T^{\ominus}$) $\left[\dfrac{\text{mol}}{\text{m}^3\,\text{Pa}}\right]$ | $\dfrac{\text{d}\ln H_s^{cp}}{\text{d}(1/T)}$ [K] | Reference | Type | Note |
|---|---|---|---|---|---|
| | $3.1\times10^{-2}$ | | Wang et al. (2017) | Q | 80, 239 |
| | $6.2\times10^{-4}$ | | Wang et al. (2017) | Q | 80, 240 |
| | $7.8\times10^{-3}$ | | Raventos-Duran et al. (2010) | Q | 242, 243 |
| | $2.0\times10^{-2}$ | | Raventos-Duran et al. (2010) | Q | 244 |
| | $3.1\times10^{-2}$ | | Raventos-Duran et al. (2010) | Q | 245 |
| | $2.5\times10^{-2}$ | | Hilal et al. (2008) | Q | |
| | $5.3\times10^{-2}$ | | Modarresi et al. (2007) | Q | 67 |
| | | 5600 | Kühne et al. (2005) | Q | |
| | $7.8\times10^{-3}$ | | Duchowicz et al. (2020) | ? | 185, 21 |
| | | 4600 | Kühne et al. (2005) | ? | |
| 2-propyl nitrate $C_3H_7ONO_2$ (isopropyl nitrate) [1712-64-7] GAPFWGOSHOCNBM-UHFFFAOYSA-N | $7.8\times10^{-3}$ | 5400 | Burkholder et al. (2019) | L | |
| | $7.8\times10^{-3}$ | 5400 | Burkholder et al. (2015) | L | |
| | $7.8\times10^{-3}$ | 5400 | Sander et al. (2011) | L | |
| | $7.8\times10^{-3}$ | 5400 | Sander et al. (2006) | L | |
| | $6.6\times10^{-3}$ | 5400 | Staudinger and Roberts (2001) | L | |
| | $5.5\times10^{-3}$ | 4300 | Hauff et al. (1998) | M | |
| | $7.8\times10^{-3}$ | 5400 | Kames and Schurath (1992) | M | |
| | $8.1\times10^{-3}$ | | Hauff et al. (1998) | V | |
| | $3.6\times10^{-3}$ | | Keshavarz et al. (2022) | Q | |
| | $2.7\times10^{-3}$ | | Duchowicz et al. (2020) | Q | 184 |
| | $1.1\times10^{-2}$ | | Wang et al. (2017) | Q | 80, 238 |
| | $2.4\times10^{-2}$ | | Wang et al. (2017) | Q | 80, 239 |
| | $7.1\times10^{-4}$ | | Wang et al. (2017) | Q | 80, 240 |
| | $7.8\times10^{-3}$ | | Raventos-Duran et al. (2010) | Q | 242, 243 |
| | $1.6\times10^{-2}$ | | Raventos-Duran et al. (2010) | Q | 244 |
| | 4.9 | | Raventos-Duran et al. (2010) | Q | 245 |
| | $1.7\times10^{-2}$ | | Hilal et al. (2008) | Q | |
| | $4.9\times10^{-2}$ | | Modarresi et al. (2007) | Q | 67 |
| | | 4600 | Kühne et al. (2005) | Q | |
| | $6.1\times10^{-3}$ | | Duchowicz et al. (2020) | ? | 185, 21 |
| | | 4300 | Kühne et al. (2005) | ? | |
| 1-butyl nitrate $C_4H_9ONO_2$ [928-45-0] QQHZPQUHCAKSOL-UHFFFAOYSA-N | $1.0\times10^{-2}$ | 5800 | Burkholder et al. (2019) | L | |
| | $1.0\times10^{-2}$ | 5800 | Burkholder et al. (2015) | L | |
| | $1.0\times10^{-2}$ | 5800 | Sander et al. (2011) | L | |
| | $1.0\times10^{-2}$ | 5800 | Sander et al. (2006) | L | |
| | $8.8\times10^{-3}$ | 6000 | Staudinger and Roberts (2001) | L | |
| | $6.3\times10^{-3}$ | 5200 | Hauff et al. (1998) | M | |
| | $1.0\times10^{-2}$ | 5800 | Kames and Schurath (1992) | M | |
| | $1.0\times10^{-2}$ | 6000 | Luke et al. (1989) | M | |
| | $8.5\times10^{-3}$ | | Hauff et al. (1998) | V | |
| | $4.8\times10^{-3}$ | | Keshavarz et al. (2022) | Q | |
| | $1.0\times10^{-2}$ | | Duchowicz et al. (2020) | Q | |
| | $8.9\times10^{-3}$ | | Wang et al. (2017) | Q | 80, 238 |
| | $2.0\times10^{-2}$ | | Wang et al. (2017) | Q | 80, 239 |
| | $3.0\times10^{-4}$ | | Wang et al. (2017) | Q | 80, 240 |
| | $4.9\times10^{-3}$ | | Raventos-Duran et al. (2010) | Q | 242, 243 |



Table A4.6: Nitrates ($RONO_2$) (... continued)

| Substance Formula (Trivial Name) [CAS Registry Number] InChIKey | $H_s^{cp}$ (at $T^{\ominus}$) $\left[\dfrac{mol}{m^3\,Pa}\right]$ | $\dfrac{d\ln H_s^{cp}}{d(1/T)}$ [K] | Reference | Type | Note |
|---|---|---|---|---|---|
| | $1.2\times10^{-2}$ | | Raventos-Duran et al. (2010) | Q | 244 |
| | $2.0\times10^{-2}$ | | Raventos-Duran et al. (2010) | Q | 245 |
| | $1.7\times10^{-2}$ | | Hilal et al. (2008) | Q | |
| | $4.2\times10^{-2}$ | | Modarresi et al. (2007) | Q | 67 |
| | | 5900 | Kühne et al. (2005) | Q | |
| | $6.4\times10^{-3}$ | | Duchowicz et al. (2020) | ? | 185, 21 |
| | | 5800 | Kühne et al. (2005) | ? | |
| 2-butyl nitrate $C_4H_9ONO_2$ [924-52-7] DYONNFFVDNILGI-UHFFFAOYSA-N | $6.4\times10^{-3}$ | 5400 | Burkholder et al. (2019) | L | |
| | $6.4\times10^{-3}$ | 5400 | Burkholder et al. (2015) | L | |
| | $6.4\times10^{-3}$ | 5400 | Sander et al. (2011) | L | |
| | $6.4\times10^{-3}$ | 5400 | Sander et al. (2006) | L | |
| | $6.4\times10^{-3}$ | 6100 | Staudinger and Roberts (2001) | L | |
| | $4.4\times10^{-3}$ | | Hauff et al. (1998) | M | |
| | $6.4\times10^{-3}$ | 5400 | Kames and Schurath (1992) | M | |
| | $6.3\times10^{-3}$ | 5600 | Luke et al. (1989) | M | |
| | $6.4\times10^{-3}$ | | Hauff et al. (1998) | V | |
| | $8.9\times10^{-3}$ | | Wang et al. (2017) | Q | 80, 238 |
| | $1.5\times10^{-2}$ | | Wang et al. (2017) | Q | 80, 239 |
| | $5.5\times10^{-4}$ | | Wang et al. (2017) | Q | 80, 240 |
| | | 4900 | Kühne et al. (2005) | Q | |
| | | 5400 | Kühne et al. (2005) | ? | |
| isobutyl nitrate $C_4H_9ONO_2$ [543-29-3] LNNXFUZKZLXPOF-UHFFFAOYSA-N | $7.0\times10^{-3}$ | 5200 | Kames and Schurath (1992) | M | |
| | $4.8\times10^{-3}$ | | Keshavarz et al. (2022) | Q | |
| | $3.9\times10^{-3}$ | | Duchowicz et al. (2020) | Q | |
| | $8.9\times10^{-3}$ | | Wang et al. (2017) | Q | 80, 238 |
| | $2.0\times10^{-2}$ | | Wang et al. (2017) | Q | 80, 239 |
| | $4.6\times10^{-4}$ | | Wang et al. (2017) | Q | 80, 240 |
| | $4.9\times10^{-3}$ | | Raventos-Duran et al. (2010) | Q | 271, 243 |
| | $9.9\times10^{-3}$ | | Raventos-Duran et al. (2010) | Q | 244 |
| | $2.0\times10^{-2}$ | | Raventos-Duran et al. (2010) | Q | 245 |
| | $1.6\times10^{-2}$ | | Hilal et al. (2008) | Q | |
| | $4.4\times10^{-2}$ | | Modarresi et al. (2007) | Q | 67 |
| | $4.4\times10^{-3}$ | | Duchowicz et al. (2020) | ? | 185, 21 |
| 1-pentyl nitrate $C_5H_{11}ONO_2$ (amyl nitrate) [1002-16-0] HSNWZBCBUUSSQD-UHFFFAOYSA-N | $6.6\times10^{-3}$ | 6300 | Hauff et al. (1998) | M | |
| | $1.2\times10^{-2}$ | | Kames and Schurath (1992) | M | 12 |
| | $4.0\times10^{-3}$ | | Hauff et al. (1998) | V | |
| | $6.5\times10^{-3}$ | | Keshavarz et al. (2022) | Q | |
| | $1.3\times10^{-2}$ | | Duchowicz et al. (2020) | Q | |
| | $6.9\times10^{-3}$ | | Wang et al. (2017) | Q | 80, 238 |
| | $1.4\times10^{-2}$ | | Wang et al. (2017) | Q | 80, 239 |
| | $4.8\times10^{-4}$ | | Wang et al. (2017) | Q | 80, 240 |
| | $3.9\times10^{-3}$ | | Raventos-Duran et al. (2010) | Q | 271, 243 |
| | $9.9\times10^{-3}$ | | Raventos-Duran et al. (2010) | Q | 244 |
| | $1.6\times10^{-2}$ | | Raventos-Duran et al. (2010) | Q | 245 |
| | $1.3\times10^{-2}$ | | Hilal et al. (2008) | Q | |
| | $3.8\times10^{-2}$ | | Modarresi et al. (2007) | Q | 67 |



Table A4.6: Nitrates ($RONO_2$) (. . . continued)

| Substance Formula (Trivial Name) [CAS Registry Number] InChIKey | $H_s^{cp}$ (at $T^\ominus$) $\left[\dfrac{\text{mol}}{\text{m}^3\,\text{Pa}}\right]$ | $\dfrac{\text{d}\ln H_s^{cp}}{\text{d}(1/T)}$ [K] | Reference | Type | Note |
|---|---|---|---|---|---|
| | | 6300 | Kühne et al. (2005) | Q | |
| | $5.9\times10^{-3}$ | | Duchowicz et al. (2020) | ? | 185, 21 |
| | | 6300 | Kühne et al. (2005) | ? | |
| 2-pentyl nitrate $C_5H_{11}ONO_2$ (MCM:PEBNO3) [21981-48-6] RWRBSYOTDDOXKC-UHFFFAOYSA-N | $3.7\times10^{-3}$ | 6400 | Staudinger and Roberts (2001) | L | |
| | $3.7\times10^{-3}$ | 5100 | Hauff et al. (1998) | M | |
| | $3.6\times10^{-3}$ | 6300 | Kames and Schurath (1992) | M | |
| | $4.8\times10^{-3}$ | | Hauff et al. (1998) | V | |
| | $3.4\times10^{-3}$ | 6000 | Wieser et al. (2023) | Q | 437 |
| | $6.5\times10^{-3}$ | | Keshavarz et al. (2022) | Q | |
| | $5.0\times10^{-3}$ | | Duchowicz et al. (2020) | Q | 299 |
| | $8.3\times10^{-3}$ | | Wang et al. (2017) | Q | 80, 238 |
| | $1.0\times10^{-2}$ | | Wang et al. (2017) | Q | 80, 239 |
| | $4.8\times10^{-4}$ | | Wang et al. (2017) | Q | 80, 240 |
| | $9.5\times10^{-3}$ | | Hilal et al. (2008) | Q | |
| | $3.6\times10^{-2}$ | | Modarresi et al. (2007) | Q | 67 |
| | | 5300 | Kühne et al. (2005) | Q | |
| | $3.3\times10^{-3}$ | | Duchowicz et al. (2020) | ? | 185, 21 |
| | | 5100 | Kühne et al. (2005) | ? | |
| 3-pentyl nitrate $C_5H_{13}ONO_2$ [82944-59-0] WQZKKVJFBZPJSU-UHFFFAOYSA-N | $3.8\times10^{-3}$ | 5300 | Hauff et al. (1998) | M | |
| | $4.9\times10^{-3}$ | | Hauff et al. (1998) | V | |
| | $6.5\times10^{-3}$ | | Keshavarz et al. (2022) | Q | |
| | $5.0\times10^{-3}$ | | Duchowicz et al. (2020) | Q | 299 |
| | $8.3\times10^{-3}$ | | Wang et al. (2017) | Q | 80, 238 |
| | $1.0\times10^{-2}$ | | Wang et al. (2017) | Q | 80, 239 |
| | $3.7\times10^{-4}$ | | Wang et al. (2017) | Q | 80, 240 |
| | $3.9\times10^{-3}$ | | Raventos-Duran et al. (2010) | Q | 242, 243 |
| | $6.2\times10^{-3}$ | | Raventos-Duran et al. (2010) | Q | 244 |
| | $1.6\times10^{-2}$ | | Raventos-Duran et al. (2010) | Q | 245 |
| | $9.2\times10^{-3}$ | | Hilal et al. (2008) | Q | |
| | $4.0\times10^{-2}$ | | Modarresi et al. (2007) | Q | 67 |
| | | 5300 | Kühne et al. (2005) | Q | |
| | $3.6\times10^{-3}$ | | Duchowicz et al. (2020) | ? | 185, 21 |
| | | 5300 | Kühne et al. (2005) | ? | |
| 3-methyl-1-butanol nitrate $C_5H_{11}ONO_2$ (isoamyl nitrate) [543-87-3] NTHGIYFSMNNHSC-UHFFFAOYSA-N | $5.0\times10^{-3}$ | 5900 | Hauff et al. (1998) | M | |
| | $6.5\times10^{-3}$ | | Keshavarz et al. (2022) | Q | |
| | $5.0\times10^{-3}$ | | Duchowicz et al. (2020) | Q | 184 |
| | $3.9\times10^{-3}$ | | Raventos-Duran et al. (2010) | Q | 271, 243 |
| | $9.9\times10^{-3}$ | | Raventos-Duran et al. (2010) | Q | 244 |
| | $1.6\times10^{-2}$ | | Raventos-Duran et al. (2010) | Q | 245 |
| | $1.2\times10^{-2}$ | | Hilal et al. (2008) | Q | |
| | $3.8\times10^{-2}$ | | Modarresi et al. (2007) | Q | 67 |
| | | 6300 | Kühne et al. (2005) | Q | |
| | $4.5\times10^{-3}$ | | Duchowicz et al. (2020) | ? | 185, 21 |
| | | 5900 | Kühne et al. (2005) | ? | |



Table A4.6: Nitrates ($RONO_2$) (...continued)

| Substance<br>Formula<br>(Trivial Name)<br>[CAS Registry Number]<br>InChIKey | $H_s^{cp}$<br>(at $T^\ominus$)<br>$\left[\dfrac{\mathrm{mol}}{\mathrm{m^3\,Pa}}\right]$ | $\dfrac{\mathrm{d}\ln H_s^{cp}}{\mathrm{d}(1/T)}$<br><br>[K] | Reference | Type | Note |
|---|---|---|---|---|---|
| pentaerythritol tetranitrate | $7.6\times10^3$ | | HSDB (2015) | V | |
| $C_5H_8N_4O_{12}$ | 1.2 | | Yaws (2003) | X | 237 |
| [78-11-5] | $8.2\times10^5$ | | Zhang et al. (2010) | Q | 287, 288 |
| TZRXHJWUDPFEEY-UHFFFAOYSA-N | $1.1\times10^4$ | | Zhang et al. (2010) | Q | 287, 289 |
| | $7.9\times10^4$ | | Zhang et al. (2010) | Q | 287, 290 |
| | $3.6\times10^3$ | | Zhang et al. (2010) | Q | 287, 291 |
| | 1.2 | | Gharagheizi et al. (2010) | Q | 246 |
| 1-hexyl nitrate | $7.6\times10^{-3}$ | 6700 | Hauff et al. (1998) | M | |
| $C_6H_{13}ONO_2$ | $3.6\times10^{-3}$ | | Hauff et al. (1998) | V | |
| [20633-11-8] | $8.8\times10^{-3}$ | | Keshavarz et al. (2022) | Q | |
| AGDYNDJUZRMYRG-UHFFFAOYSA-N | $1.5\times10^{-2}$ | | Duchowicz et al. (2020) | Q | 184 |
| | $5.8\times10^{-3}$ | | Wang et al. (2017) | Q | 80, 238 |
| | $1.0\times10^{-2}$ | | Wang et al. (2017) | Q | 80, 239 |
| | $2.4\times10^{-4}$ | | Wang et al. (2017) | Q | 80, 240 |
| | $3.1\times10^{-3}$ | | Raventos-Duran et al. (2010) | Q | 271, 243 |
| | $6.2\times10^{-3}$ | | Raventos-Duran et al. (2010) | Q | 244 |
| | $1.2\times10^{-2}$ | | Raventos-Duran et al. (2010) | Q | 245 |
| | $9.5\times10^{-3}$ | | Hilal et al. (2008) | Q | |
| | $3.1\times10^{-2}$ | | Modarresi et al. (2007) | Q | 67 |
| | | 6600 | Kühne et al. (2005) | Q | |
| | $6.6\times10^{-3}$ | | Duchowicz et al. (2020) | ? | 185, 21 |
| | | 6700 | Kühne et al. (2005) | ? | |
| 2-nitrooxyethanol | $3.9\times10^2$ | | Burkholder et al. (2019) | L | |
| $HOC_2H_4ONO_2$ | $3.9\times10^2$ | | Burkholder et al. (2015) | L | |
| [16051-48-2] | $3.9\times10^2$ | | Sander et al. (2011) | L | |
| HTKIMWYSDZQQBP-UHFFFAOYSA-N | $3.9\times10^2$ | | Sander et al. (2006) | L | |
| | $3.8\times10^2$ | 8600 | Shepson et al. (1996) | M | |
| | $3.9\times10^2$ | | Kames and Schurath (1992) | M | 12 |
| | $5.7\times10^1$ | | Keshavarz et al. (2022) | Q | |
| | $9.0\times10^{-1}$ | | Duchowicz et al. (2020) | Q | 299 |
| | $4.2\times10^1$ | | Wang et al. (2017) | Q | 80, 238 |
| | $5.5\times10^2$ | | Wang et al. (2017) | Q | 80, 239 |
| | 7.6 | | Wang et al. (2017) | Q | 80, 240 |
| | $3.9\times10^1$ | | Raventos-Duran et al. (2010) | Q | 242, 243 |
| | $2.0\times10^2$ | | Raventos-Duran et al. (2010) | Q | 244 |
| | $9.9\times10^2$ | | Raventos-Duran et al. (2010) | Q | 245 |
| | $1.7\times10^2$ | | Hilal et al. (2008) | Q | |
| | $8.5\times10^1$ | | Modarresi et al. (2007) | Q | 67 |
| | | 9200 | Kühne et al. (2005) | Q | |
| | $3.8\times10^2$ | | Duchowicz et al. (2020) | ? | 185, 21 |
| | | 8700 | Kühne et al. (2005) | ? | |



Table A4.6: Nitrates ($RONO_2$) (... continued)

| Substance Formula (Trivial Name) [CAS Registry Number] InChIKey | $H_s^{cp}$ (at $T^\ominus$) $\left[\dfrac{\mathrm{mol}}{\mathrm{m^3\,Pa}}\right]$ | $\dfrac{\mathrm{d}\ln H_s^{cp}}{\mathrm{d}(1/T)}$ [K] | Reference | Type | Note |
|---|---|---|---|---|---|
| 1-nitrooxy-2-propanol $C_3H_7O_4N$ [20266-65-3] OMDFJTSKBNDDRM-UHFFFAOYSA-N | $6.6\times10^1$ | | Burkholder et al. (2019) | L | |
| | $6.6\times10^1$ | | Burkholder et al. (2015) | L | |
| | $6.6\times10^1$ | | Sander et al. (2011) | L | |
| | $6.6\times10^1$ | | Sander et al. (2006) | L | |
| | $1.1\times10^2$ | 10000 | Shepson et al. (1996) | M | |
| | $6.6\times10^1$ | | Kames and Schurath (1992) | M | 578, 12 |
| | $7.2\times10^1$ | | Kames and Schurath (1992) | M | 578, 12 |
| | $7.7\times10^1$ | | Keshavarz et al. (2022) | Q | |
| | $9.3\times10^{-1}$ | | Duchowicz et al. (2020) | Q | |
| | $3.9\times10^1$ | | Wang et al. (2017) | Q | 80, 238 |
| | $3.6\times10^2$ | | Wang et al. (2017) | Q | 80, 239 |
| | 1.7 | | Wang et al. (2017) | Q | 80, 240 |
| | $3.1\times10^1$ | | Raventos-Duran et al. (2010) | Q | 242, 243 |
| | $1.2\times10^2$ | | Raventos-Duran et al. (2010) | Q | 244 |
| | $7.8\times10^2$ | | Raventos-Duran et al. (2010) | Q | 245 |
| | $9.5\times10^1$ | | Hilal et al. (2008) | Q | |
| | $5.5\times10^1$ | | Modarresi et al. (2007) | Q | 67 |
| | $1.1\times10^2$ | | Duchowicz et al. (2020) | ? | 185, 21 |
| 2-nitrooxy-1-propanol $C_3H_7O_4N$ [20266-74-4] HGCMMKIIGJXXMW-UHFFFAOYSA-N | $7.2\times10^1$ | | Burkholder et al. (2019) | L | |
| | $7.2\times10^1$ | | Burkholder et al. (2015) | L | |
| | $7.2\times10^1$ | | Sander et al. (2011) | L | |
| | $7.2\times10^1$ | | Sander et al. (2006) | L | |
| | $4.4\times10^1$ | 8800 | Shepson et al. (1996) | M | |
| | $6.6\times10^1$ | | Kames and Schurath (1992) | M | 578, 12 |
| | $7.2\times10^1$ | | Kames and Schurath (1992) | M | 578, 12 |
| | $7.7\times10^1$ | | Keshavarz et al. (2022) | Q | |
| | $9.3\times10^{-1}$ | | Duchowicz et al. (2020) | Q | 184 |
| | $3.9\times10^1$ | | Wang et al. (2017) | Q | 80, 238 |
| | $2.7\times10^2$ | | Wang et al. (2017) | Q | 80, 239 |
| | 1.6 | | Wang et al. (2017) | Q | 80, 240 |
| | $3.1\times10^1$ | | Raventos-Duran et al. (2010) | Q | 242, 243 |
| | $9.9\times10^1$ | | Raventos-Duran et al. (2010) | Q | 244 |
| | $7.8\times10^2$ | | Raventos-Duran et al. (2010) | Q | 245 |
| | $8.6\times10^1$ | | Hilal et al. (2008) | Q | |
| | $6.9\times10^1$ | | Modarresi et al. (2007) | Q | 67 |
| | $4.4\times10^1$ | | Duchowicz et al. (2020) | ? | 185, 21 |
| 1-nitrooxy-2-butanol $C_4H_9O_4N$ [147794-11-4] KNUQGVIXAYDSOX-UHFFFAOYSA-N | $8.9\times10^1$ | | Treves et al. (2000) | M | 28 |
| | $5.7\times10^1$ | 9200 | Shepson et al. (1996) | M | |
| | $1.0\times10^2$ | | Keshavarz et al. (2022) | Q | |
| | 1.7 | | Duchowicz et al. (2020) | Q | 299 |
| | $3.2\times10^1$ | | Wang et al. (2017) | Q | 80, 238 |
| | $1.9\times10^2$ | | Wang et al. (2017) | Q | 80, 239 |
| | 1.1 | | Wang et al. (2017) | Q | 80, 240 |
| | $2.5\times10^1$ | | Raventos-Duran et al. (2010) | Q | 242, 243 |
| | $7.8\times10^1$ | | Raventos-Duran et al. (2010) | Q | 244 |
| | $6.2\times10^2$ | | Raventos-Duran et al. (2010) | Q | 245 |





Table A4.6: Nitrates ($RONO_2$) (. . . continued)

| Substance<br>Formula<br>(Trivial Name)<br>[CAS Registry Number]<br>InChIKey | $H_s^{cp}$<br>(at $T^\ominus$)<br>$\left[\dfrac{\text{mol}}{\text{m}^3\,\text{Pa}}\right]$ | $\dfrac{\text{d}\ln H_s^{cp}}{\text{d}(1/T)}$<br><br>[K] | Reference | Type | Note |
|---|---|---|---|---|---|
| | $6.1\times10^1$ | | Hilal et al. (2008) | Q | |
| | $5.4\times10^1$ | | Modarresi et al. (2007) | Q | 67 |
| | $5.7\times10^1$ | | Duchowicz et al. (2020) | ? | 185, 21 |
| 2-nitrooxy-1-butanol | $8.8\times10^1$ | | Treves et al. (2000) | M | 28 |
| $C_4H_9O_4N$ | $5.9\times10^1$ | 9600 | Shepson et al. (1996) | M | |
| [147794-12-5] | $1.0\times10^2$ | | Keshavarz et al. (2022) | Q | |
| YXMNEYKMHSBVTD-UHFFFAOYSA-N | 1.7 | | Duchowicz et al. (2020) | Q | 299 |
| | $3.2\times10^1$ | | Wang et al. (2017) | Q | 80, 238 |
| | $1.6\times10^2$ | | Wang et al. (2017) | Q | 80, 239 |
| | 1.6 | | Wang et al. (2017) | Q | 80, 240 |
| | $2.5\times10^1$ | | Raventos-Duran et al. (2010) | Q | 242, 243 |
| | $6.2\times10^1$ | | Raventos-Duran et al. (2010) | Q | 244 |
| | $6.2\times10^2$ | | Raventos-Duran et al. (2010) | Q | 245 |
| | $6.0\times10^1$ | | Hilal et al. (2008) | Q | |
| | $5.4\times10^1$ | | Modarresi et al. (2007) | Q | 67 |
| | $5.9\times10^1$ | | Duchowicz et al. (2020) | ? | 185, 21 |
| 2-nitrooxy-3-butanol | $1.0\times10^2$ | 9500 | Shepson et al. (1996) | M | |
| $C_4H_9O_4N$ | $1.0\times10^2$ | | Keshavarz et al. (2022) | Q | |
| [147794-10-3] | $6.8\times10^{-1}$ | | Duchowicz et al. (2020) | Q | 299 |
| CGFCSKMZZXPWEY-UHFFFAOYSA-N | $3.6\times10^1$ | | Wang et al. (2017) | Q | 80, 238 |
| | $2.0\times10^2$ | | Wang et al. (2017) | Q | 80, 239 |
| | $9.3\times10^{-1}$ | | Wang et al. (2017) | Q | 80, 240 |
| | $2.5\times10^1$ | | Raventos-Duran et al. (2010) | Q | 271, 243 |
| | $7.8\times10^1$ | | Raventos-Duran et al. (2010) | Q | 244 |
| | $6.2\times10^2$ | | Raventos-Duran et al. (2010) | Q | 245 |
| | $5.4\times10^1$ | | Hilal et al. (2008) | Q | |
| | $7.4\times10^1$ | | Modarresi et al. (2007) | Q | 67 |
| | $1.0\times10^2$ | | Duchowicz et al. (2020) | ? | 185, 21 |
| 3-nitrooxy-1-butanol | $1.4\times10^2$ | | Treves et al. (2000) | M | 28 |
| $C_4H_9O_4N$ | $2.6\times10^1$ | | Wang et al. (2017) | Q | 80, 238 |
| FOHXKGXDMSQVTH-UHFFFAOYSA-N | $5.1\times10^2$ | | Wang et al. (2017) | Q | 80, 239 |
| | 5.0 | | Wang et al. (2017) | Q | 80, 240 |
| 4-nitrooxy-1-butanol | $2.9\times10^2$ | | Treves et al. (2000) | M | 28 |
| $C_4H_9O_4N$ | $2.2\times10^1$ | | Wang et al. (2017) | Q | 80, 238 |
| [22911-39-3] | $7.4\times10^2$ | | Wang et al. (2017) | Q | 80, 239 |
| FBOGSWRRYABFKU-UHFFFAOYSA-N | $1.6\times10^1$ | | Wang et al. (2017) | Q | 80, 240 |
| 4-nitrooxy-2-butanol | $1.3\times10^2$ | | Treves et al. (2000) | M | 28 |
| $C_4H_9O_4N$ | $2.6\times10^1$ | | Wang et al. (2017) | Q | 80, 238 |
| (3-hydroxy-1-nitrooxy-butane) | $6.5\times10^2$ | | Wang et al. (2017) | Q | 80, 239 |
| [141299-18-5] | 3.4 | | Wang et al. (2017) | Q | 80, 240 |
| WUKDMTQXKGFHBU-UHFFFAOYSA-N | | | | | |





Table A4.6: Nitrates ($RONO_2$) (. . . continued)

| Substance Formula (Trivial Name) [CAS Registry Number] InChIKey | $H_s^{cp}$ (at $T^\ominus$) $\left[\dfrac{\text{mol}}{\text{m}^3\,\text{Pa}}\right]$ | $\dfrac{\mathrm{d}\ln H_s^{cp}}{\mathrm{d}(1/T)}$ [K] | Reference | Type | Note |
|---|---|---|---|---|---|
| 4-nitrooxy-1-pentanol $C_5H_{11}O_4N$ SVIFKHUNGQLAEN-UHFFFAOYSA-N | $2.0\times10^2$ $2.0\times10^1$ $3.4\times10^2$ $1.3\times10^1$ | | Treves et al. (2000) Wang et al. (2017) Wang et al. (2017) Wang et al. (2017) | M Q Q Q | 28 80, 238 80, 239 80, 240 |
| 5-nitrooxy-1-pentanol $C_5H_{11}O_4N$ DVOSRLSVEOCANS-UHFFFAOYSA-N | $2.0\times10^2$ | | Ebert et al. (2023) | ? | 579 |
| 5-nitrooxy-2-pentanol $C_5H_{11}O_4N$ (MCM:HO2C5NO3) RIQPKERROQFFJK-UHFFFAOYSA-N | $3.6\times10^2$ $2.1\times10^1$ $2.0\times10^1$ $4.4\times10^2$ $6.5$ | 9900 | Treves et al. (2000) Wieser et al. (2023) Wang et al. (2017) Wang et al. (2017) Wang et al. (2017) | M Q Q Q Q | 28 437 80, 238 80, 239 80, 240 |
| 1-nitrooxy-2-propanone $CH_3COCH_2ONO_2$ (nitrooxyacetone) [6745-71-7] ISWXYJQANHQYSR-UHFFFAOYSA-N | $1.0\times10^1$ $1.0\times10^1$ $1.0\times10^1$ $1.0\times10^1$ $1.0\times10^1$ $7.6$ $1.4\times10^2$ $5.0\times10^{-1}$ $2.5\times10^1$ $3.9\times10^1$ $7.8\times10^1$ $1.2\times10^2$ | | Burkholder et al. (2019) Burkholder et al. (2015) Sander et al. (2011) Sander et al. (2006) Kames and Schurath (1992) Wang et al. (2017) Wang et al. (2017) Wang et al. (2017) Raventos-Duran et al. (2010) Raventos-Duran et al. (2010) Raventos-Duran et al. (2010) Hilal et al. (2008) | L L L L M Q Q Q Q Q Q Q | 12 80, 238 80, 239 80, 240 242, 243 244 245 |
| ISOP1N6CO $C_5H_7NO_5$ XXORYENTKCFFSM-UHFFFAOYSA-N | $2.1\times10^5$ | 14000 | Wieser et al. (2023) | Q | 437 |
| C52COCONO2 $C_5H_7NO_5$ QJXLMYNMVOSKND-UHFFFAOYSA-N | $6.7\times10^1$ | 11000 | Wieser et al. (2023) | Q | 437 |
| ISOP1N23O4CO $C_5H_7NO_5$ WBVYILWVTRBTDJ-UHFFFAOYSA-N | $1.5\times10^3$ | 12000 | Wieser et al. (2023) | Q | 437 |
| ROO6R7ONO2 $C_6H_{11}NO_5$ COKJDYWOSQONMU-UHFFFAOYSA-N | $7.5\times10^4$ | 13000 | Wieser et al. (2023) | Q | 437 |
| 2-heptyl nitrate $C_7H_{15}NO_3$ (C7H15ONO2) HHXLSUKHLTZWKR-UHFFFAOYSA-N | $2.3\times10^{-3}$ | 6600 | Wieser et al. (2023) | Q | 437 |



Table A4.6: Nitrates ($RONO_2$) (...continued)

| Substance Formula (Trivial Name) [CAS Registry Number] InChIKey | $H_s^{cp}$ (at $T^\ominus$) $\left[\dfrac{\text{mol}}{\text{m}^3\,\text{Pa}}\right]$ | $\dfrac{\text{d}\ln H_s^{cp}}{\text{d}(1/T)}$ [K] | Reference | Type | Note |
|---|---|---|---|---|---|
| C7OHONO2 $C_7H_{15}NO_4$ MZJHMUSTUUJGSJ-UHFFFAOYSA-N | $1.2\times10^1$ | 11000 | Wieser et al. (2023) | Q | 437 |
| C622CONO2 $C_7H_{11}NO_5$ DJCVHBPUJHAWIU-UHFFFAOYSA-N | $9.9\times10^2$ | 14000 | Wieser et al. (2023) | Q | 437 |
| 2-octyl nitrate $C_8H_{17}NO_3$ (C8H17ONO2) QCOKASLKYUXYJH-UHFFFAOYSA-N | 6.8 | 7000 | Wieser et al. (2023) | Q | 437 |
| C8OHONO2 $C_8H_{17}NO_4$ UEQWMROPIQKFIM-UHFFFAOYSA-N | 9.0 | 11000 | Wieser et al. (2023) | Q | 437 |
| C824ONO2 $C_8H_{13}NO_5$ YMYILPYDFNRVBZ-UHFFFAOYSA-N | $9.5\times10^6$ | 14000 | Wieser et al. (2023) | Q | 437 |
| C819ONO2 $C_8H_{13}NO_6$ XLUWXWLWQTYVOF-UHFFFAOYSA-N | $3.9\times10^6$ | 16000 | Wieser et al. (2023) | Q | 437 |
| C92ONO2 $C_9H_{15}NO_5$ PNFDSIMYWKOAKE-UHFFFAOYSA-N | $4.7\times10^5$ | 15000 | Wieser et al. (2023) | Q | 437 |
| NORLIMONO2 $C_9H_{15}NO_6$ MUTWZANDXPDWFF-UHFFFAOYSA-N | $3.8\times10^8$ | 16000 | Wieser et al. (2023) | Q | 437 |
| C822CONO2 $C_9H_{13}NO_5$ RVFLVBTUEWAABB-UHFFFAOYSA-N | $6.2\times10^1$ | 13000 | Wieser et al. (2023) | Q | 437 |
| C817CONO2 $C_9H_{13}NO_6$ LHFXXFGIVAQFTO-UHFFFAOYSA-N | $1.2\times10^5$ | 15000 | Wieser et al. (2023) | Q | 437 |
| C9CONO2 $C_{10}H_{15}NO_5$ FNVMVXZLPDKVAS-UHFFFAOYSA-N | $8.0\times10^1$ | 20000 | Wieser et al. (2023) | Q | 437 |
| LIMONO2 $C_{10}H_{15}NO_5$ DEKWHDOXDJNWRD-UHFFFAOYSA-N | $3.8\times10^4$ | 14000 | Wieser et al. (2023) | Q | 437 |



Table A4.6: Nitrates ($RONO_2$) (...continued)

| Substance<br>Formula<br>(Trivial Name)<br>[CAS Registry Number]<br><small>InChIKey</small> | $H_s^{cp}$<br>(at $T^\ominus$)<br>$\left[\dfrac{\text{mol}}{\text{m}^3\,\text{Pa}}\right]$ | $\dfrac{\text{d}\ln H_s^{cp}}{\text{d}(1/T)}$<br><br>[K] | Reference | Type | Note |
|---|---|---|---|---|---|
| 1,2-ethanediol dinitrate | 6.3 | | Burkholder et al. (2019) | L | |
| $O_3NCH_2CH_2ONO_2$ | 6.3 | | Burkholder et al. (2015) | L | |
| (1,2-ethane dinitrate) | 6.3 | | Sander et al. (2011) | L | |
| [628-96-6] | 6.3 | | Sander et al. (2006) | L | |
| UQXKXGWGFRWILX-UHFFFAOYSA-N | $7.8\times10^{-1}$ | | Fischer and Ballschmiter (1998b) | M | 580 |
| | 6.3 | | Kames and Schurath (1992) | M | 12 |
| | $9.9\times10^{-1}$ | | Raventos-Duran et al. (2010) | Q | 271, 243 |
| | 9.9 | | Raventos-Duran et al. (2010) | Q | 244 |
| | $1.6\times10^{1}$ | | Raventos-Duran et al. (2010) | Q | 245 |
| | 8.2 | | Hilal et al. (2008) | Q | |
| | $4.9\times10^{-1}$ | | Modarresi et al. (2007) | Q | 67 |
| 1,2-propanediol dinitrate | 1.7 | | Burkholder et al. (2019) | L | |
| $C_3H_6(ONO_2)_2$ | 1.7 | | Burkholder et al. (2015) | L | |
| (1,2-propane dinitrate) | 1.7 | | Sander et al. (2011) | L | |
| [6423-43-4] | 1.7 | | Sander et al. (2006) | L | |
| PSXCGTLGGVDWFU-UHFFFAOYSA-N | $3.2\times10^{-1}$ | | Fischer and Ballschmiter (1998b) | M | 580 |
| | 1.7 | | Kames and Schurath (1992) | M | 12 |
| | $1.0\times10^{1}$ | | HSDB (2015) | Q | 99 |
| | $7.8\times10^{-1}$ | | Raventos-Duran et al. (2010) | Q | 271, 243 |
| | 3.1 | | Raventos-Duran et al. (2010) | Q | 244 |
| | 9.9 | | Raventos-Duran et al. (2010) | Q | 245 |
| | 2.7 | | Hilal et al. (2008) | Q | |
| 1,3-propanediol dinitrate | 1.6 | | Burkholder et al. (2019) | L | 581 |
| $C_3H_6N_2O_6$ | 1.6 | | Burkholder et al. (2015) | L | 582 |
| [3457-90-7] | 1.3 | | Fischer and Ballschmiter (1998b) | M | 580 |
| KOSAMXZBGUIISK-UHFFFAOYSA-N | 1.2 | | Raventos-Duran et al. (2010) | Q | 242, 243 |
| | 3.9 | | Raventos-Duran et al. (2010) | Q | 244 |
| | 9.9 | | Raventos-Duran et al. (2010) | Q | 245 |
| | 4.4 | | Hilal et al. (2008) | Q | |
| 1,2,3-propanetriol trinitrate | $2.3\times10^{2}$ | | HSDB (2015) | V | |
| $C_3H_5N_3O_9$ | $1.9\times10^{2}$ | | Yaws (2003) | X | 237, 12 |
| (nitroglycerin) | $4.9\times10^{1}$ | | Raventos-Duran et al. (2010) | Q | 242, 243 |
| [55-63-0] | $3.9\times10^{2}$ | | Raventos-Duran et al. (2010) | Q | 244 |
| SNIOPGDIGTZGOP-UHFFFAOYSA-N | $3.9\times10^{3}$ | | Raventos-Duran et al. (2010) | Q | 245 |
| | $1.9\times10^{2}$ | | Gharagheizi et al. (2010) | Q | 246 |
| | $3.9\times10^{1}$ | | Hilal et al. (2008) | Q | |
| | $1.0\times10^{2}$ | | Yaws (1999) | ? | 21, 12 |
| 1,2-butanediol dinitrate | $2.1\times10^{-1}$ | | Fischer and Ballschmiter (1998b) | M | 580 |
| $C_4H_8N_2O_6$ | $4.9\times10^{-1}$ | | Raventos-Duran et al. (2010) | Q | 271, 243 |
| [20820-41-1] | 1.6 | | Raventos-Duran et al. (2010) | Q | 244 |
| CTISQZXTUUHJNC-UHFFFAOYSA-N | 7.8 | | Raventos-Duran et al. (2010) | Q | 245 |



Table A4.6: Nitrates ($RONO_2$) (...continued)

| Substance<br>Formula<br>(Trivial Name)<br>[CAS Registry Number]<br>InChIKey | $H_s^{cp}$<br>(at $T^\ominus$)<br>$\left[ \dfrac{\mathrm{mol}}{\mathrm{m^3\,Pa}} \right]$ | $\dfrac{\mathrm{d}\ln H_s^{cp}}{\mathrm{d}(1/T)}$<br><br>[K] | Reference | Type | Note |
|---|---|---|---|---|---|
| 1,3-butanediol dinitrate<br>$C_4H_8N_2O_6$<br>[6423-44-5]<br>DGFBULNMARLFTH-UHFFFAOYSA-N | $5.7\times10^{-1}$<br>$9.9\times10^{-1}$<br>1.6<br>7.8 | | Fischer and Ballschmiter (1998b)<br>Raventos-Duran et al. (2010)<br>Raventos-Duran et al. (2010)<br>Raventos-Duran et al. (2010) | M<br>Q<br>Q<br>Q | 580<br>242, 243<br>244<br>245 |
| 1,4-butanediol dinitrate<br>$C_4H_8N_2O_6$<br>[3457-91-8]<br>QELUAJBXJAWSRC-UHFFFAOYSA-N | 1.6<br>$9.9\times10^{-1}$<br>3.1<br>7.8<br>2.7 | | Fischer and Ballschmiter (1998b)<br>Raventos-Duran et al. (2010)<br>Raventos-Duran et al. (2010)<br>Raventos-Duran et al. (2010)<br>Hilal et al. (2008) | M<br>Q<br>Q<br>Q<br>Q | 580<br>242, 243<br>244<br>245 |
| 2,3-butanediol dinitrate<br>$C_4H_8N_2O_6$<br>[6423-45-6]<br>RVDDYBGRQLZMSB-UHFFFAOYSA-N | $1.2\times10^{-1}$<br>$4.9\times10^{-1}$<br>1.6<br>7.8 | | Fischer and Ballschmiter (1998b)<br>Raventos-Duran et al. (2010)<br>Raventos-Duran et al. (2010)<br>Raventos-Duran et al. (2010) | M<br>Q<br>Q<br>Q | 580<br>242, 243<br>244<br>245 |
| 1,2-pentanediol dinitrate<br>$C_5H_{10}N_2O_6$<br>[89365-05-9]<br>MZWHVGLDAWPHHW-UHFFFAOYSA-N | $1.3\times10^{-1}$ | | Fischer and Ballschmiter (1998b) | M | 580 |
| 1,4-pentanediol dinitrate<br>$C_5H_{10}N_2O_6$<br>[25385-63-1]<br>IUTIKUKYGRINOD-UHFFFAOYSA-N | $3.9\times10^{-1}$ | | Fischer and Ballschmiter (1998b) | M | 580 |
| 1,5-pentanediol dinitrate<br>$C_5H_{10}N_2O_6$<br>[3457-92-9]<br>MIYIEPHJPVBSEV-UHFFFAOYSA-N | 1.2<br>$9.9\times10^{-1}$<br>2.0<br>6.2 | | Fischer and Ballschmiter (1998b)<br>Raventos-Duran et al. (2010)<br>Raventos-Duran et al. (2010)<br>Raventos-Duran et al. (2010) | M<br>Q<br>Q<br>Q | 580<br>242, 243<br>244<br>245 |
| (2R,4S)-2,4-pentanediol dinitrate<br>$C_5H_{10}N_2O_6$<br>(*cis*-2,4-pentanediol dinitrate)<br>[208252-05-5] | $2.2\times10^{-1}$ | | Fischer and Ballschmiter (1998b) | M | 580 |
| (2R,4R)-2,4-pentanediol dinitrate<br>$C_5H_{10}N_2O_6$<br>(*trans*-2,4-pentanediol dinitrate)<br>[208252-04-4] | $1.4\times10^{-1}$ | | Fischer and Ballschmiter (1998b) | M | 580 |
| 1,2-hexanediol dinitrate<br>$C_6H_{12}N_2O_6$<br>[110539-07-6]<br>UJKJGCZXZPXTGS-UHFFFAOYSA-N | $9.6\times10^{-2}$ | | Fischer and Ballschmiter (1998b) | M | 580 |
| 1,5-hexanediol dinitrate<br>$C_6H_{12}N_2O_6$<br>[206443-83-6]<br>PGDWEAOSOZNKFU-UHFFFAOYSA-N | $2.7\times10^{-1}$ | | Fischer and Ballschmiter (1998b) | M | 580 |



Table A4.6: Nitrates ($RONO_2$) (... continued)

| Substance<br>Formula<br>(Trivial Name)<br>[CAS Registry Number]<br>InChIKey | $H_s^{cp}$ (at $T^\ominus$)<br><br>$\left[\dfrac{\mathrm{mol}}{\mathrm{m^3\,Pa}}\right]$ | $\dfrac{\mathrm{d}\ln H_s^{cp}}{\mathrm{d}(1/T)}$<br><br>[K] | Reference | Type | Note |
|---|---|---|---|---|---|
| 1,6-hexanediol dinitrate<br>$C_6H_{12}N_2O_6$<br>[3457-93-0]<br>GCVAYIWGKFFWEU-UHFFFAOYSA-N | 1.5 | | Fischer and Ballschmiter (1998b) | M | 580 |
| 2,5-hexanediol dinitrate<br>$C_6H_{12}N_2O_6$<br>[99115-63-6]<br>ISSLCMSTXXUOEU-UHFFFAOYSA-N | $3.1\times10^{-1}$<br>$6.2\times10^{-1}$<br>$6.2\times10^{-1}$<br>4.9 | | Fischer and Ballschmiter (1998b)<br>Raventos-Duran et al. (2010)<br>Raventos-Duran et al. (2010)<br>Raventos-Duran et al. (2010) | M<br>Q<br>Q<br>Q | 580<br>271, 243<br>244<br>245 |
| (1R,2S)-1,2-cyclohexanediol dinitrate<br>$C_6H_{10}N_2O_6$<br>(*cis*-1,2-cyclohexanediol dinitrate)<br>[32342-28-2] | 1.3 | | Fischer and Ballschmiter (1998b) | M | 580 |
| (1R,2R)-1,2-cyclohexanediol dinitrate<br>$C_6H_{10}N_2O_6$<br>(*trans*-1,2-cyclohexanediol dinitrate)<br>[32342-29-3] | $5.2\times10^{-1}$ | | Fischer and Ballschmiter (1998b) | M | 580 |
| (1R,3S)-1,3-cyclohexanediol dinitrate<br>$C_6H_{10}N_2O_6$<br>(*cis*-1,3-cyclohexanediol dinitrate)<br>[170994-36-2] | 3.4 | | Fischer and Ballschmiter (1998b) | M | 580 |
| (1R,3R)-1,3-cyclohexanediol dinitrate<br>$C_6H_{10}N_2O_6$<br>(*trans*-1,3-cyclohexanediol dinitrate)<br>[170994-41-9] | $6.8\times10^{-1}$ | | Fischer and Ballschmiter (1998b) | M | 580 |
| 1,7-heptanediol dinitrate<br>$C_7H_{14}N_2O_6$<br>[3457-94-1]<br>KIERETFMVSIXIJ-UHFFFAOYSA-N | 1.1 | | Fischer and Ballschmiter (1998b) | M | 580 |
| (1R,2R)-1,2-cycloheptanediol dinitrate<br>$C_7H_{12}N_2O_6$<br>(*trans*-1,2-cycloheptanediol dinitrate)<br>[208252-06-6] | $8.8\times10^{-1}$ | | Fischer and Ballschmiter (1998b) | M | 580 |





Table A4.6: Nitrates ($RONO_2$) (...continued)

| Substance<br>Formula<br>(Trivial Name)<br>[CAS Registry Number]<br>InChIKey | $H_s^{cp}$<br>(at $T^{\ominus}$)<br>$\left[\dfrac{\text{mol}}{\text{m}^3\,\text{Pa}}\right]$ | $\dfrac{\text{d}\ln H_s^{cp}}{\text{d}(1/T)}$<br><br>[K] | Reference | Type | Note |
|---|---|---|---|---|---|
| 1,2-octanediol dinitrate<br>$C_8H_{16}N_2O_6$<br>[121222-48-8]<br>RLNKMZWMCSTIEM-UHFFFAOYSA-N | $5.2\times10^{-2}$ | | Fischer and Ballschmiter (1998b) | M | 580 |
| 1,8-octanediol dinitrate<br>$C_8H_{16}N_2O_6$<br>[3457-95-2]<br>BVIHOKFRPSZJEI-UHFFFAOYSA-N | $7.8\times10^{-1}$ | | Fischer and Ballschmiter (1998b) | M | 580 |
| 1,2-decanediol dinitrate<br>$C_{10}H_{20}N_2O_6$<br>[60123-40-2]<br>KKNGVXOJDIUADE-UHFFFAOYSA-N | $2.0\times10^{-2}$ | | Fischer and Ballschmiter (1998b) | M | 580 |
| 1,10-decanediol dinitrate<br>$C_{10}H_{20}N_2O_6$<br>[3457-97-4]<br>RHUZOYMELURURD-UHFFFAOYSA-N | $4.3\times10^{-1}$ | | Fischer and Ballschmiter (1998b) | M | 580 |
| diethylene glycol dinitrate<br>$C_4H_8N_2O_7$<br>[693-21-0]<br>LYAGTVMJGHTIDH-UHFFFAOYSA-N | $2.5\times10^{1}$<br>$4.9\times10^{1}$<br>$1.2\times10^{2}$<br>$9.9\times10^{2}$<br>$1.1\times10^{2}$<br>$3.8$ | | HSDB (2015)<br>Raventos-Duran et al. (2010)<br>Raventos-Duran et al. (2010)<br>Raventos-Duran et al. (2010)<br>Hilal et al. (2008)<br>Modarresi et al. (2007) | V<br>Q<br>Q<br>Q<br>Q<br>Q | <br>242, 243<br>244<br>245<br><br>67 |
| ISOP1N2ONO2<br>$C_5H_8N_2O_6$<br>FQSKJUXVWIFNKX-UHFFFAOYSA-N | $1.6$ | 9400 | Wieser et al. (2023) | Q | 437 |
| EPXISOPNONO2<br>$C_5H_8N_2O_7$<br>UAFKAIYDADHRMX-UHFFFAOYSA-N | $5.3\times10^{1}$ | 13000 | Wieser et al. (2023) | Q | 437 |
| ISOP1N23O4ONO2<br>$C_5H_8N_2O_7$<br>FSZUIXLMCRZJMS-UHFFFAOYSA-N | $4.5\times10^{1}$ | 13000 | Wieser et al. (2023) | Q | 437 |
| ISOP1N5ONO2<br>$C_5H_8N_2O_7$<br>WRGRJJOKHABEJT-UHFFFAOYSA-N | $5.8\times10^{4}$ | 13000 | Wieser et al. (2023) | Q | 437 |
| LIMAB15ONO22<br>$C_{10}H_{18}N_2O_8$<br>UNHNJOVXNACOSQ-UHFFFAOYSA-N | $3.1\times10^{7}$ | 18000 | Wieser et al. (2023) | Q | 437 |



Table A4.6: Nitrates ($RONO_2$) (...continued)

| Substance Formula (Trivial Name) [CAS Registry Number] InChIKey | $H_s^{cp}$ (at $T^\ominus$) $\left[\dfrac{\text{mol}}{\text{m}^3\,\text{Pa}}\right]$ | $\dfrac{\text{d}\ln H_s^{cp}}{\text{d}(1/T)}$ [K] | Reference | Type | Note |
|---|---|---|---|---|---|
| peroxyacetyl nitrate | $2.8\times10^{-2}$ | 5700 | Burkholder et al. (2019) | L | |
| $CH_3COOONO_2$ | $2.8\times10^{-2}$ | 5700 | Burkholder et al. (2015) | L | |
| (PAN) | $2.9\times10^{-2}$ | 5700 | Warneck and Williams (2012) | L | |
| [2278-22-0] | $2.8\times10^{-2}$ | 5700 | Sander et al. (2011) | L | |
| VGQXTTSVLMQFHM-UHFFFAOYSA-N | $2.8\times10^{-2}$ | 5700 | Sander et al. (2006) | L | |
| | $2.9\times10^{-2}$ | 5800 | Leu and Zhang (1999) | L | |
| | $3.0\times10^{-2}$ | 5600 | Easterbrook et al. (2023) | M | |
| | $2.3\times10^{-2}$ | 4800 | Frenzel et al. (2000) | M | |
| | $4.0\times10^{-2}$ | | Kames and Schurath (1995) | M | 12 |
| | $2.8\times10^{-2}$ | 6500 | Kames et al. (1991) | M | |
| | $4.9\times10^{-2}$ | | Holdren et al. (1984) | M | 373 |
| | $3.6\times10^{-2}$ | | Gaffney and Senum (1984) | X | 389 |
| | $2.9\times10^{-2}$ | 5900 | Pandis and Seinfeld (1989) | C | |
| | $3.6\times10^{-2}$ | | Schwartz (1986) | C | 87 |
| | $6.8\times10^{-2}$ | | Keshavarz et al. (2022) | Q | |
| | $6.1\times10^{-1}$ | | Duchowicz et al. (2020) | Q | 184 |
| | 4.9 | | Wang et al. (2017) | Q | 80, 238 |
| | $1.0\times10^{1}$ | | Wang et al. (2017) | Q | 80, 239 |
| | $7.4\times10^{-4}$ | | Wang et al. (2017) | Q | 80, 240 |
| | $3.1\times10^{-2}$ | | Raventos-Duran et al. (2010) | Q | 271, 243 |
| | 4.9 | | Raventos-Duran et al. (2010) | Q | 244 |
| | $7.8\times10^{-2}$ | | Raventos-Duran et al. (2010) | Q | 245 |
| | 2.2 | | Hilal et al. (2008) | Q | |
| | 3.2 | | Modarresi et al. (2007) | Q | 67 |
| | | 4800 | Kühne et al. (2005) | Q | |
| | $3.6\times10^{-2}$ | | Duchowicz et al. (2020) | ? | 185, 21 |
| | | 6300 | Kühne et al. (2005) | ? | |
| | | | Warneck et al. (1996) | ? | 583 |
| | | | Schurath et al. (1996) | W | 584 |
| peroxypropionyl nitrate | $1.6\times10^{-2}$ | 6000 | Easterbrook et al. (2023) | M | |
| $C_2H_5COOONO_2$ | $2.9\times10^{-2}$ | | Kames and Schurath (1995) | M | 12 |
| (PPN) | 3.9 | | Wang et al. (2017) | Q | 80, 238 |
| [5796-89-4] | 4.1 | | Wang et al. (2017) | Q | 80, 239 |
| TXINBPKSWKFMNB-UHFFFAOYSA-N | $3.6\times10^{-4}$ | | Wang et al. (2017) | Q | 80, 240 |
| | $2.5\times10^{-2}$ | | Raventos-Duran et al. (2010) | Q | 242, 243 |
| | 2.0 | | Raventos-Duran et al. (2010) | Q | 244 |
| | $6.2\times10^{-2}$ | | Raventos-Duran et al. (2010) | Q | 245 |
| | | | Warneck et al. (1996) | ? | 583 |
| | | | Schurath et al. (1996) | W | 585 |
| nitro butaneperoxoate | $2.3\times10^{-2}$ | | Kames and Schurath (1995) | M | 12 |
| $C_3H_7COOONO_2$ | 3.2 | | Wang et al. (2017) | Q | 80, 238 |
| (PnBN) | 2.2 | | Wang et al. (2017) | Q | 80, 239 |
| [27746-48-1] | $2.6\times10^{-4}$ | | Wang et al. (2017) | Q | 80, 240 |
| HZUMMZVMNQSPFF-UHFFFAOYSA-N | $1.6\times10^{-2}$ | | Raventos-Duran et al. (2010) | Q | 242, 243 |
| | 1.2 | | Raventos-Duran et al. (2010) | Q | 244 |
| | $4.9\times10^{-2}$ | | Raventos-Duran et al. (2010) | Q | 245 |



Table A4.6: Nitrates ($RONO_2$) (... continued)

| Substance Formula (Trivial Name) [CAS Registry Number] InChIKey | $H_s^{cp}$ (at $T^\ominus$) $\left[\dfrac{\text{mol}}{\text{m}^3\,\text{Pa}}\right]$ | $\dfrac{\text{d}\ln H_s^{cp}}{\text{d}(1/T)}$ [K] | Reference | Type | Note |
|---|---|---|---|---|---|
| | | | Warneck et al. (1996) | ? | 583 |
| | | | Schurath et al. (1996) | W | 586 |
| peroxy-2-propenoyl nitrate | $1.7\times10^{-2}$ | | Kames and Schurath (1995) | M | 12 |
| $CH_2C(CH_3)COOONO_2$ | 7.4 | | Wang et al. (2017) | Q | 80, 238 |
| (peroxymethacryloyl nitrate; | 4.8 | | Wang et al. (2017) | Q | 80, 239 |
| MPAN) | | | | | |
| [88181-75-3] | $2.4\times10^{-4}$ | | Wang et al. (2017) | Q | 80, 240 |
| LLZWPQFQEBKRLX-UHFFFAOYSA-N | | | Warneck et al. (1996) | W | 583 |
| | | | Schurath et al. (1996) | W | 587 |
| peroxy-isobutyryl nitrate | $9.9\times10^{-3}$ | | Kames and Schurath (1995) | M | 12 |
| $C_3H_7COOONO_2$ | 3.6 | | Wang et al. (2017) | Q | 80, 238 |
| (PiBN) | 2.3 | | Wang et al. (2017) | Q | 80, 239 |
| [65424-60-4] | $1.8\times10^{-4}$ | | Wang et al. (2017) | Q | 80, 240 |
| BDNFHGUXBRZLRQ-UHFFFAOYSA-N | $1.6\times10^{-2}$ | | Raventos-Duran et al. (2010) | Q | 242, 243 |
| | 1.2 | | Raventos-Duran et al. (2010) | Q | 244 |
| | $4.9\times10^{-2}$ | | Raventos-Duran et al. (2010) | Q | 245 |
| | | | Warneck et al. (1996) | ? | 583 |
| | | | Schurath et al. (1996) | W | 588 |
| furoyl peroxynitrate | $9.3\times10^{-2}$ | 8800 | Roberts et al. (2022) | M | |
| $C_5H_3NO_6$ | $1.6\times10^{-1}$ | | Roberts et al. (2022) | Q | 589 |
| (fur-PAN) | | | | | |
| NMNZVYLGOWCHOD-UHFFFAOYSA-N | | | | | |
| MCM:CH3O2NO2 | $4.1\times10^{-1}$ | | Wang et al. (2017) | Q | 80, 238 |
| $CH_3NO_4$ | $1.5\times10^{-1}$ | | Wang et al. (2017) | Q | 80, 239 |
| LCFGXMPUQSXLCQ-UHFFFAOYSA-N | $1.7\times10^{-4}$ | | Wang et al. (2017) | Q | 80, 240 |
| MCM:ETHO2HNO3 | $9.3\times10^2$ | | Wang et al. (2017) | Q | 80, 238 |
| $C_2H_5NO_5$ | $2.8\times10^3$ | | Wang et al. (2017) | Q | 80, 239 |
| IYCQDYYQELIXMU-UHFFFAOYSA-N | 3.1 | | Wang et al. (2017) | Q | 80, 240 |
| MCM:NO3CH2CO3H | $1.2\times10^4$ | | Wang et al. (2017) | Q | 80, 238 |
| $C_2H_3NO_6$ | $2.3\times10^3$ | | Wang et al. (2017) | Q | 80, 239 |
| VLCOOHYMNNDETJ-UHFFFAOYSA-N | $4.4\times10^{-1}$ | | Wang et al. (2017) | Q | 80, 240 |
| MCM:NO3CH2PAN | $5.6\times10^2$ | | Wang et al. (2017) | Q | 80, 238 |
| $C_2H_2N_2O_8$ | $1.9\times10^3$ | | Wang et al. (2017) | Q | 80, 239 |
| CSNUPQYBRZIENU-UHFFFAOYSA-N | $3.0\times10^{-4}$ | | Wang et al. (2017) | Q | 80, 240 |
| MCM:ACRPAN | $1.1\times10^1$ | | Wang et al. (2017) | Q | 80, 238 |
| $C_3H_3NO_5$ | 7.4 | | Wang et al. (2017) | Q | 80, 239 |
| [157258-66-7] | $4.6\times10^{-4}$ | | Wang et al. (2017) | Q | 80, 240 |
| SFKRQQJZRXQGLC-UHFFFAOYSA-N | $2.0\times10^{-2}$ | | Raventos-Duran et al. (2010) | Q | 271, 243 |
| | 3.1 | | Raventos-Duran et al. (2010) | Q | 244 |
| | $1.6\times10^{-1}$ | | Raventos-Duran et al. (2010) | Q | 245 |



Table A4.6: Nitrates ($RONO_2$) (...continued)

| Substance Formula (Trivial Name) [CAS Registry Number] InChIKey | $H_s^{cp}$ (at $T^{\ominus}$) $\left[\dfrac{\text{mol}}{\text{m}^3\,\text{Pa}}\right]$ | $\dfrac{\text{d}\ln H_s^{cp}}{\text{d}(1/T)}$ [K] | Reference | Type | Note |
|---|---|---|---|---|---|
| MCM:PR1O2HNO3 | $8.7\times10^2$ | | Wang et al. (2017) | Q | 80, 238 |
| $C_3H_7NO_5$ | $1.0\times10^3$ | | Wang et al. (2017) | Q | 80, 239 |
| FROUFTMXOYPVCX-UHFFFAOYSA-N | 1.5 | | Wang et al. (2017) | Q | 80, 240 |
| MCM:PR2O2HNO3 | $8.7\times10^2$ | | Wang et al. (2017) | Q | 80, 238 |
| $C_3H_7NO_5$ | $1.1\times10^3$ | | Wang et al. (2017) | Q | 80, 239 |
| UHIVGIHVGRYUPL-UHFFFAOYSA-N | 1.7 | | Wang et al. (2017) | Q | 80, 240 |
| MCM:PRNO3CO3H | $1.1\times10^4$ | | Wang et al. (2017) | Q | 80, 238 |
| $C_3H_5NO_6$ | $1.0\times10^3$ | | Wang et al. (2017) | Q | 80, 239 |
| FMRKJWQAPYMRJD-UHFFFAOYSA-N | 1.3 | | Wang et al. (2017) | Q | 80, 240 |
| MCM:PRNO3PAN | $5.0\times10^2$ | | Wang et al. (2017) | Q | 80, 238 |
| $C_3H_4N_2O_8$ | $4.6\times10^2$ | | Wang et al. (2017) | Q | 80, 239 |
| ZLJYZUGEJFPONP-UHFFFAOYSA-N | $1.6\times10^{-4}$ | | Wang et al. (2017) | Q | 80, 240 |
| MCM:BU1ENO3OOH | $6.8\times10^2$ | | Wang et al. (2017) | Q | 80, 238 |
| $C_4H_9NO_5$ | $5.5\times10^2$ | | Wang et al. (2017) | Q | 80, 239 |
| MFTWJXGCASCYHP-UHFFFAOYSA-N | 1.0 | | Wang et al. (2017) | Q | 80, 240 |
| MCM:C3DBPAN | $1.4\times10^1$ | | Wang et al. (2017) | Q | 80, 238 |
| $C_4H_5NO_5$ | 5.8 | | Wang et al. (2017) | Q | 80, 239 |
| YVSVRLGWMDVSPN-UHFFFAOYSA-N | $1.2\times10^{-3}$ | | Wang et al. (2017) | Q | 80, 240 |
| MCM:C42NO33OOH | $8.1\times10^2$ | | Wang et al. (2017) | Q | 80, 238 |
| $C_4H_9NO_5$ | $5.1\times10^2$ | | Wang et al. (2017) | Q | 80, 239 |
| CGBCWMNQUOIZIB-UHFFFAOYSA-N | $9.6\times10^{-1}$ | | Wang et al. (2017) | Q | 80, 240 |
| MCM:C43NO34OOH | $6.8\times10^2$ | | Wang et al. (2017) | Q | 80, 238 |
| $C_4H_9NO_5$ | $5.5\times10^2$ | | Wang et al. (2017) | Q | 80, 239 |
| BQDGWSCZHPCGGT-UHFFFAOYSA-N | $6.0\times10^{-1}$ | | Wang et al. (2017) | Q | 80, 240 |
| MCM:C4PAN9 | $1.6\times10^3$ | | Wang et al. (2017) | Q | 80, 238 |
| $C_4H_4N_2O_8$ | $2.3\times10^3$ | | Wang et al. (2017) | Q | 80, 239 |
| SHASFEZZYJNCIM-UHFFFAOYSA-N | $5.4\times10^{-3}$ | | Wang et al. (2017) | Q | 80, 240 |
| MCM:MPRANO3OOH | $4.7\times10^2$ | | Wang et al. (2017) | Q | 80, 238 |
| $C_4H_9NO_5$ | $3.2\times10^2$ | | Wang et al. (2017) | Q | 80, 239 |
| DTPMQEMACNJTBJ-UHFFFAOYSA-N | $9.6\times10^{-1}$ | | Wang et al. (2017) | Q | 80, 240 |
| MCM:MPRBNO3OOH | $4.7\times10^2$ | | Wang et al. (2017) | Q | 80, 238 |
| $C_4H_9NO_5$ | $3.5\times10^2$ | | Wang et al. (2017) | Q | 80, 239 |
| RCQWCNCMOFDCKD-UHFFFAOYSA-N | 5.1 | | Wang et al. (2017) | Q | 80, 240 |
| MCM:MPRBNO3PAN | $2.9\times10^2$ | | Wang et al. (2017) | Q | 80, 238 |
| $C_4H_6N_2O_8$ | $1.1\times10^2$ | | Wang et al. (2017) | Q | 80, 239 |
| RAYRXKOIFAVDJK-UHFFFAOYSA-N | $2.1\times10^{-4}$ | | Wang et al. (2017) | Q | 80, 240 |
| MCM:MPRNO3CO3H | $6.2\times10^3$ | | Wang et al. (2017) | Q | 80, 238 |
| $C_4H_7NO_6$ | $2.2\times10^2$ | | Wang et al. (2017) | Q | 80, 239 |
| HDNCPEGDTKOFQI-UHFFFAOYSA-N | $5.8\times10^{-1}$ | | Wang et al. (2017) | Q | 80, 240 |



Table A4.6: Nitrates ($RONO_2$) (. . . continued)

| Substance Formula (Trivial Name) [CAS Registry Number] InChIKey | $H_s^{cp}$ (at $T^\ominus$) $\left[\dfrac{\text{mol}}{\text{m}^3\,\text{Pa}}\right]$ | $\dfrac{\text{d}\ln H_s^{cp}}{\text{d}(1/T)}$ [K] | Reference | Type | Note |
|---|---|---|---|---|---|
| MCM:NBUTDAOOH $C_4H_7NO_5$ ZIAMBHNGSMXNCM-UHFFFAOYSA-N | $2.0\times10^3$ $1.2\times10^3$ $6.3$ | | Wang et al. (2017) Wang et al. (2017) Wang et al. (2017) | Q Q Q | 80, 238 80, 239 80, 240 |
| MCM:NBUTDBNO3 $C_4H_6N_2O_6$ GCQJZAFTGYZGDG-UHFFFAOYSA-N | $4.1$ $3.6\times10^1$ $1.4\times10^{-2}$ | | Wang et al. (2017) Wang et al. (2017) Wang et al. (2017) | Q Q Q | 80, 238 80, 239 80, 240 |
| MCM:NBUTDBOOH $C_4H_7NO_5$ SSNXKZNERJCDKO-UHFFFAOYSA-N | $2.8\times10^3$ $4.0\times10^3$ $5.6\times10^1$ | | Wang et al. (2017) Wang et al. (2017) Wang et al. (2017) | Q Q Q | 80, 238 80, 239 80, 240 |
| MCM:NC3CO3H $C_4H_5NO_6$ ZXNVOGOFZJSFAE-UHFFFAOYSA-N | $3.6\times10^4$ $5.0\times10^3$ $3.7\times10^{-1}$ | | Wang et al. (2017) Wang et al. (2017) Wang et al. (2017) | Q Q Q | 80, 238 80, 239 80, 240 |
| MCM:TC4H9NO3 $C_4H_9NO_3$ [926-05-6] AZAKMLHUDVIDFN-UHFFFAOYSA-N | $6.2\times10^{-3}$ $9.1\times10^{-3}$ $1.0\times10^{-3}$ $4.9\times10^{-3}$ $6.2\times10^{-3}$ $2.0\times10^{-2}$ | | Wang et al. (2017) Wang et al. (2017) Wang et al. (2017) Raventos-Duran et al. (2010) Raventos-Duran et al. (2010) Raventos-Duran et al. (2010) | Q Q Q Q Q Q | 80, 238 80, 239 80, 240 242, 243 244 245 |
| MCM:C3ME3PAN $C_5H_9NO_5$ OLNWAQLSTCXURT-UHFFFAOYSA-N | $2.8$ $1.7$ $1.6\times10^{-4}$ | | Wang et al. (2017) Wang et al. (2017) Wang et al. (2017) | Q Q Q | 80, 238 80, 239 80, 240 |
| MCM:C3MNO3CO3H $C_5H_9NO_6$ UNTXQXBLSROXLI-UHFFFAOYSA-N | $9.3\times10^3$ $3.2\times10^2$ $2.7\times10^{-1}$ | | Wang et al. (2017) Wang et al. (2017) Wang et al. (2017) | Q Q Q | 80, 238 80, 239 80, 240 |
| MCM:C3MNO3PAN $C_5H_8N_2O_8$ ULZBZSVLLIBTGC-UHFFFAOYSA-N | $3.9\times10^2$ $1.6\times10^2$ $1.0\times10^{-4}$ | | Wang et al. (2017) Wang et al. (2017) Wang et al. (2017) | Q Q Q | 80, 238 80, 239 80, 240 |
| MCM:C43NO3CO3H $C_5H_9NO_6$ KYFRJEXZVMOAMN-UHFFFAOYSA-N | $5.8\times10^3$ $1.3\times10^2$ $1.6\times10^{-2}$ | | Wang et al. (2017) Wang et al. (2017) Wang et al. (2017) | Q Q Q | 80, 238 80, 239 80, 240 |
| MCM:C43NO3PAN $C_5H_8N_2O_8$ WPFKHDFVFPXDFZ-UHFFFAOYSA-N | $2.2\times10^2$ $6.6\times10^1$ $1.3\times10^{-4}$ | | Wang et al. (2017) Wang et al. (2017) Wang et al. (2017) | Q Q Q | 80, 238 80, 239 80, 240 |
| MCM:C4M2NO3OOH $C_5H_{11}NO_5$ BOAAVKBCMOPNIQ-UHFFFAOYSA-N | $4.5\times10^2$ $1.6\times10^2$ $5.8\times10^{-1}$ | | Wang et al. (2017) Wang et al. (2017) Wang et al. (2017) | Q Q Q | 80, 238 80, 239 80, 240 |
| MCM:C4MNO31OOH $C_5H_{11}NO_5$ LRRBWUKXCGVENZ-UHFFFAOYSA-N | $6.3\times10^2$ $3.2\times10^2$ $7.3\times10^{-1}$ | | Wang et al. (2017) Wang et al. (2017) Wang et al. (2017) | Q Q Q | 80, 238 80, 239 80, 240 |





Table A4.6: Nitrates ($RONO_2$) (...continued)

| Substance<br>Formula<br>(Trivial Name)<br>[CAS Registry Number]<br>InChIKey | $H_s^{cp}$<br>(at $T^{\ominus}$)<br>$\left[\dfrac{\text{mol}}{\text{m}^3\,\text{Pa}}\right]$ | $\dfrac{\text{d}\ln H_s^{cp}}{\text{d}(1/T)}$<br><br>[K] | Reference | Type | Note |
|---|---|---|---|---|---|
| MCM:C4MNO32OOH<br>$C_5H_{11}NO_5$<br>HXNZNKRZULBVRO-UHFFFAOYSA-N | $6.3\times10^2$<br>$3.2\times10^2$<br>$4.7\times10^{-1}$ | | Wang et al. (2017)<br>Wang et al. (2017)<br>Wang et al. (2017) | Q<br>Q<br>Q | 80, 238<br>80, 239<br>80, 240 |
| MCM:C4NO32MOOH<br>$C_5H_{11}NO_5$<br>HCKTYRRBEFAEAC-UHFFFAOYSA-N | $3.9\times10^2$<br>$1.7\times10^2$<br>$5.6\times10^{-1}$ | | Wang et al. (2017)<br>Wang et al. (2017)<br>Wang et al. (2017) | Q<br>Q<br>Q | 80, 238<br>80, 239<br>80, 240 |
| MCM:C4NO3CO3H<br>$C_5H_9NO_6$<br>JPUVYTOGMBAUBQ-UHFFFAOYSA-N | $8.3\times10^3$<br>$3.0\times10^2$<br>$1.7\times10^{-2}$ | | Wang et al. (2017)<br>Wang et al. (2017)<br>Wang et al. (2017) | Q<br>Q<br>Q | 80, 238<br>80, 239<br>80, 240 |
| MCM:C4NO3M1OOH<br>$C_5H_{11}NO_5$<br>MWSZSODDMCBKIX-UHFFFAOYSA-N | $3.9\times10^2$<br>$1.8\times10^2$<br>$3.4$ | | Wang et al. (2017)<br>Wang et al. (2017)<br>Wang et al. (2017) | Q<br>Q<br>Q | 80, 238<br>80, 239<br>80, 240 |
| MCM:C4NO3M2OOH<br>$C_5H_{11}NO_5$<br>QXHHHGYKLMZJBA-UHFFFAOYSA-N | $4.5\times10^2$<br>$1.5\times10^2$<br>$4.2$ | | Wang et al. (2017)<br>Wang et al. (2017)<br>Wang et al. (2017) | Q<br>Q<br>Q | 80, 238<br>80, 239<br>80, 240 |
| MCM:C4NO3PAN<br>$C_5H_8N_2O_8$<br>GGZFCBHNGNBLNW-UHFFFAOYSA-N | $3.2\times10^2$<br>$1.6\times10^2$<br>$1.2\times10^{-4}$ | | Wang et al. (2017)<br>Wang et al. (2017)<br>Wang et al. (2017) | Q<br>Q<br>Q | 80, 238<br>80, 239<br>80, 240 |
| MCM:C51NO32OOH<br>$C_5H_{11}NO_5$<br>YPLQKABNDDXYJG-UHFFFAOYSA-N | $6.3\times10^2$<br>$3.2\times10^2$<br>$1.1$ | | Wang et al. (2017)<br>Wang et al. (2017)<br>Wang et al. (2017) | Q<br>Q<br>Q | 80, 238<br>80, 239<br>80, 240 |
| MCM:C52NO31OOH<br>$C_5H_{11}NO_5$<br>DCAGTGOPLFTTTQ-UHFFFAOYSA-N | $6.3\times10^2$<br>$3.2\times10^2$<br>$4.3\times10^{-1}$ | | Wang et al. (2017)<br>Wang et al. (2017)<br>Wang et al. (2017) | Q<br>Q<br>Q | 80, 238<br>80, 239<br>80, 240 |
| MCM:C52NO33OOH<br>$C_5H_{11}NO_5$<br>FQGADFCNSPWVJM-UHFFFAOYSA-N | $6.3\times10^2$<br>$2.4\times10^2$<br>$4.2\times10^{-1}$ | | Wang et al. (2017)<br>Wang et al. (2017)<br>Wang et al. (2017) | Q<br>Q<br>Q | 80, 238<br>80, 239<br>80, 240 |
| MCM:C53NO32OOH<br>$C_5H_{11}NO_5$<br>STNFTACLRXEYDM-UHFFFAOYSA-N | $6.3\times10^2$<br>$2.6\times10^2$<br>$2.3\times10^{-1}$ | | Wang et al. (2017)<br>Wang et al. (2017)<br>Wang et al. (2017) | Q<br>Q<br>Q | 80, 238<br>80, 239<br>80, 240 |
| MCM:C5PAN18<br>$C_5H_6N_2O_8$<br>JYUXMKSXMIBDSN-UHFFFAOYSA-N | $1.1\times10^3$<br>$2.2\times10^3$<br>$2.0\times10^{-3}$ | | Wang et al. (2017)<br>Wang et al. (2017)<br>Wang et al. (2017) | Q<br>Q<br>Q | 80, 238<br>80, 239<br>80, 240 |
| MCM:C5PAN4<br>$C_5H_9NO_5$<br>LGNOWHJLTKBAFC-UHFFFAOYSA-N | $2.8$<br>$1.5$<br>$1.3\times10^{-4}$ | | Wang et al. (2017)<br>Wang et al. (2017)<br>Wang et al. (2017) | Q<br>Q<br>Q | 80, 238<br>80, 239<br>80, 240 |
| MCM:C5PAN6<br>$C_5H_9NO_5$<br>VQRIYOXAZWZCOV-UHFFFAOYSA-N | $2.0$<br>$1.4$<br>$1.6\times10^{-4}$ | | Wang et al. (2017)<br>Wang et al. (2017)<br>Wang et al. (2017) | Q<br>Q<br>Q | 80, 238<br>80, 239<br>80, 240 |





Table A4.6: Nitrates ($RONO_2$) (...continued)

| Substance<br>Formula<br>(Trivial Name)<br>[CAS Registry Number]<br>InChIKey | $H_s^{cp}$<br>(at $T^{\ominus}$)<br>$\left[\dfrac{mol}{m^3\,Pa}\right]$ | $\dfrac{d\ln H_s^{cp}}{d(1/T)}$<br>[K] | Reference | Type | Note |
|---|---|---|---|---|---|
| MCM:IPEANO3<br>$C_5H_{11}NO_3$<br>QSSWLPYLLKHHLB-UHFFFAOYSA-N | $8.3\times10^{-3}$<br>$1.5\times10^{-2}$<br>$4.0\times10^{-4}$ | | Wang et al. (2017)<br>Wang et al. (2017)<br>Wang et al. (2017) | Q<br>Q<br>Q | 80, 238<br>80, 239<br>80, 240 |
| MCM:IPEBNO3<br>$C_5H_{11}NO_3$<br>OTVLXFGCWJFXJU-UHFFFAOYSA-N | $8.3\times10^{-3}$<br>$1.4\times10^{-2}$<br>$4.2\times10^{-4}$ | | Wang et al. (2017)<br>Wang et al. (2017)<br>Wang et al. (2017) | Q<br>Q<br>Q | 80, 238<br>80, 239<br>80, 240 |
| MCM:IPECNO3<br>$C_5H_{11}NO_3$<br>UENFRVTUGZKXNH-UHFFFAOYSA-N | $5.1\times10^{-3}$<br>$6.6\times10^{-3}$<br>$7.6\times10^{-4}$ | | Wang et al. (2017)<br>Wang et al. (2017)<br>Wang et al. (2017) | Q<br>Q<br>Q | 80, 238<br>80, 239<br>80, 240 |
| MCM:NC4CO3H<br>$C_5H_7NO_6$<br>SLKHOLOIOGBFQB-UHFFFAOYSA-N | $2.3\times10^{4}$<br>$4.6\times10^{3}$<br>$1.7\times10^{-1}$ | | Wang et al. (2017)<br>Wang et al. (2017)<br>Wang et al. (2017) | Q<br>Q<br>Q | 80, 238<br>80, 239<br>80, 240 |
| MCM:NEOPNO3<br>$C_5H_{11}NO_3$<br>YJGBGWFCCCQXIM-UHFFFAOYSA-N | $5.1\times10^{-3}$<br>$1.2\times10^{-2}$<br>$4.1\times10^{-4}$ | | Wang et al. (2017)<br>Wang et al. (2017)<br>Wang et al. (2017) | Q<br>Q<br>Q | 80, 238<br>80, 239<br>80, 240 |
| MCM:NISOPNO3<br>$C_5H_8N_2O_6$<br>JGJBVRGABXKDRR-UHFFFAOYSA-N | 2.6<br>2.7<br>$3.0\times10^{1}$<br>$1.3\times10^{-2}$ | 9400 | Wieser et al. (2023)<br>Wang et al. (2017)<br>Wang et al. (2017)<br>Wang et al. (2017) | Q<br>Q<br>Q<br>Q | 437<br>80, 238<br>80, 239<br>80, 240 |
| MCM:NISOPOOH<br>$C_5H_9NO_5$<br>IRFXVIPRKCCSLU-UHFFFAOYSA-N | $9.7\times10^{2}$<br>$1.8\times10^{3}$<br>$3.0\times10^{3}$<br>$3.0\times10^{1}$ | 11000 | Wieser et al. (2023)<br>Wang et al. (2017)<br>Wang et al. (2017)<br>Wang et al. (2017) | Q<br>Q<br>Q<br>Q | 437<br>80, 238<br>80, 239<br>80, 240 |
| MCM:PPEN<br>$C_5H_9NO_5$<br>UUYNBJNAIGBJAO-UHFFFAOYSA-N | 2.8<br>1.5<br>$2.2\times10^{-4}$ | | Wang et al. (2017)<br>Wang et al. (2017)<br>Wang et al. (2017) | Q<br>Q<br>Q | 80, 238<br>80, 239<br>80, 240 |
| MCM:C42MNO3OOH<br>$C_6H_{13}NO_5$<br>KKTQLMUUOWWZBO-UHFFFAOYSA-N | $2.4\times10^{2}$<br>$4.5\times10^{1}$<br>1.0 | | Wang et al. (2017)<br>Wang et al. (2017)<br>Wang et al. (2017) | Q<br>Q<br>Q | 80, 238<br>80, 239<br>80, 240 |
| MCM:C62NO33OOH<br>$C_6H_{13}NO_5$<br>PJRWVMFJYMZPCW-UHFFFAOYSA-N | $5.6\times10^{2}$<br>$1.5\times10^{2}$<br>$4.2\times10^{-1}$ | | Wang et al. (2017)<br>Wang et al. (2017)<br>Wang et al. (2017) | Q<br>Q<br>Q | 80, 238<br>80, 239<br>80, 240 |
| MCM:C63NO32OOH<br>$C_6H_{13}NO_5$<br>RLIDOIUHNSZVBK-UHFFFAOYSA-N | $5.6\times10^{2}$<br>$1.9\times10^{2}$<br>$1.7\times10^{-1}$ | | Wang et al. (2017)<br>Wang et al. (2017)<br>Wang et al. (2017) | Q<br>Q<br>Q | 80, 238<br>80, 239<br>80, 240 |
| MCM:C65NO36OOH<br>$C_6H_{13}NO_5$<br>ZVRSMGCQHDCSBC-UHFFFAOYSA-N | $4.9\times10^{2}$<br>$2.2\times10^{2}$<br>$2.2\times10^{-1}$ | | Wang et al. (2017)<br>Wang et al. (2017)<br>Wang et al. (2017) | Q<br>Q<br>Q | 80, 238<br>80, 239<br>80, 240 |
| MCM:C65NO3CO3H<br>$C_6H_{11}NO_6$<br>RASQUKWKKRVIQJ-UHFFFAOYSA-N | $6.5\times10^{3}$<br>$2.2\times10^{2}$<br>$1.4\times10^{-2}$ | | Wang et al. (2017)<br>Wang et al. (2017)<br>Wang et al. (2017) | Q<br>Q<br>Q | 80, 238<br>80, 239<br>80, 240 |





Table A4.6: Nitrates ($RONO_2$) (... continued)

| Substance<br>Formula<br>(Trivial Name)<br>[CAS Registry Number]<br>InChIKey | $H_s^{cp}$<br>(at $T^{\ominus}$)<br>$\left[\dfrac{\mathrm{mol}}{\mathrm{m^3\,Pa}}\right]$ | $\dfrac{\mathrm{d}\ln H_s^{cp}}{\mathrm{d}(1/T)}$<br><br>[K] | Reference | Type | Note |
|---|---|---|---|---|---|
| MCM:C65NO3PAN<br>$C_6H_{10}N_2O_8$<br>PNNYZMYYAAWSJH-UHFFFAOYSA-N | $3.0\times10^2$<br>$1.1\times10^2$<br>$6.5\times10^{-5}$ | | Wang et al. (2017)<br>Wang et al. (2017)<br>Wang et al. (2017) | Q<br>Q<br>Q | 80, 238<br>80, 239<br>80, 240 |
| MCM:C66NO35OOH<br>$C_6H_{13}NO_5$<br>YPVNGNKLOFDKIF-UHFFFAOYSA-N | $4.9\times10^2$<br>$2.3\times10^2$<br>$9.3\times10^{-1}$ | | Wang et al. (2017)<br>Wang et al. (2017)<br>Wang et al. (2017) | Q<br>Q<br>Q | 80, 238<br>80, 239<br>80, 240 |
| MCM:C6PAN10<br>$C_6H_{11}NO_5$<br>INNSDEOCGQAUTR-UHFFFAOYSA-N | $2.6$<br>$1.3$<br>$1.8\times10^{-4}$ | | Wang et al. (2017)<br>Wang et al. (2017)<br>Wang et al. (2017) | Q<br>Q<br>Q | 80, 238<br>80, 239<br>80, 240 |
| MCM:C6PAN15<br>$C_6H_{11}NO_5$<br>GWYKJYISUPTXQD-UHFFFAOYSA-N | $1.6$<br>$9.8\times10^{-1}$<br>$1.4\times10^{-4}$ | | Wang et al. (2017)<br>Wang et al. (2017)<br>Wang et al. (2017) | Q<br>Q<br>Q | 80, 238<br>80, 239<br>80, 240 |
| MCM:C6PAN17<br>$C_6H_{11}NO_5$<br>CUMLTNLXWSMQLS-UHFFFAOYSA-N | $1.6$<br>$1.1$<br>$1.7\times10^{-4}$ | | Wang et al. (2017)<br>Wang et al. (2017)<br>Wang et al. (2017) | Q<br>Q<br>Q | 80, 238<br>80, 239<br>80, 240 |
| MCM:C6PAN18<br>$C_6H_{11}NO_5$<br>FPWBUKNMBKRQCI-UHFFFAOYSA-N | $3.0$<br>$1.4$<br>$1.4\times10^{-4}$ | | Wang et al. (2017)<br>Wang et al. (2017)<br>Wang et al. (2017) | Q<br>Q<br>Q | 80, 238<br>80, 239<br>80, 240 |
| MCM:C6PAN3<br>$C_6H_{11}NO_5$<br>BRVOFEXYGAZHOW-UHFFFAOYSA-N | $2.6$<br>$1.1$<br>$1.5\times10^{-4}$ | | Wang et al. (2017)<br>Wang et al. (2017)<br>Wang et al. (2017) | Q<br>Q<br>Q | 80, 238<br>80, 239<br>80, 240 |
| MCM:CHEXNO3<br>$C_6H_{11}NO_3$<br>HLYOOCIMLHNMOG-UHFFFAOYSA-N | $2.0\times10^{-2}$<br>$6.5\times10^{-2}$<br>$3.7\times10^{-3}$ | | Wang et al. (2017)<br>Wang et al. (2017)<br>Wang et al. (2017) | Q<br>Q<br>Q | 80, 238<br>80, 239<br>80, 240 |
| MCM:HEXBNO3<br>$C_6H_{13}NO_3$<br>JLGBOJJVQLNMGV-UHFFFAOYSA-N | $3.1\times10^{-4}$<br>$6.5\times10^{-3}$<br>$7.4\times10^{-3}$<br>$4.3\times10^{-4}$ | $6300$ | Wieser et al. (2023)<br>Wang et al. (2017)<br>Wang et al. (2017)<br>Wang et al. (2017) | Q<br>Q<br>Q<br>Q | 437<br>80, 238<br>80, 239<br>80, 240 |
| MCM:HEXCNO3<br>$C_6H_{13}NO_3$<br>OJOZCOXRANAOPV-UHFFFAOYSA-N | $6.5\times10^{-3}$<br>$8.1\times10^{-3}$<br>$3.2\times10^{-4}$ | | Wang et al. (2017)<br>Wang et al. (2017)<br>Wang et al. (2017) | Q<br>Q<br>Q | 80, 238<br>80, 239<br>80, 240 |
| MCM:M22C43NO3<br>$C_6H_{13}NO_3$<br>YUZOHIKIQPTSGJ-UHFFFAOYSA-N | $4.8\times10^{-3}$<br>$9.3\times10^{-3}$<br>$3.7\times10^{-4}$ | | Wang et al. (2017)<br>Wang et al. (2017)<br>Wang et al. (2017) | Q<br>Q<br>Q | 80, 238<br>80, 239<br>80, 240 |
| MCM:M22C4NO3<br>$C_6H_{13}NO_3$<br>JOWPFKXUGRKJFA-UHFFFAOYSA-N | $4.6\times10^{-3}$<br>$1.0\times10^{-2}$<br>$5.5\times10^{-4}$ | | Wang et al. (2017)<br>Wang et al. (2017)<br>Wang et al. (2017) | Q<br>Q<br>Q | 80, 238<br>80, 239<br>80, 240 |
| MCM:M23C43NO3<br>$C_6H_{13}NO_3$<br>PMABLOPIFVACLQ-UHFFFAOYSA-N | $4.8\times10^{-3}$<br>$7.4\times10^{-3}$<br>$7.4\times10^{-4}$ | | Wang et al. (2017)<br>Wang et al. (2017)<br>Wang et al. (2017) | Q<br>Q<br>Q | 80, 238<br>80, 239<br>80, 240 |





Table A4.6: Nitrates ($RONO_2$) (...continued)

| Substance<br>Formula<br>(Trivial Name)<br>[CAS Registry Number]<br>InChIKey | $H_s^{cp}$<br>(at $T^\ominus$)<br>$\left[\dfrac{\text{mol}}{\text{m}^3\,\text{Pa}}\right]$ | $\dfrac{\text{d}\ln H_s^{cp}}{\text{d}(1/T)}$<br><br>[K] | Reference | Type | Note |
|---|---|---|---|---|---|
| MCM:M23C4NO3<br>$C_6H_{13}NO_3$<br>HHRFTVZVEUIWEO-UHFFFAOYSA-N | $7.8\times10^{-3}$<br>$1.5\times10^{-2}$<br>$4.4\times10^{-4}$ | | Wang et al. (2017)<br>Wang et al. (2017)<br>Wang et al. (2017) | Q<br>Q<br>Q | 80, 238<br>80, 239<br>80, 240 |
| MCM:M2PEANO3<br>$C_6H_{13}NO_3$<br>CETUNBADQVCCRS-UHFFFAOYSA-N | $6.5\times10^{-3}$<br>$1.1\times10^{-2}$<br>$3.4\times10^{-4}$ | | Wang et al. (2017)<br>Wang et al. (2017)<br>Wang et al. (2017) | Q<br>Q<br>Q | 80, 238<br>80, 239<br>80, 240 |
| MCM:M2PEBNO3<br>$C_6H_{13}NO_3$<br>AYWPQHSAUPNICG-UHFFFAOYSA-N | $7.8\times10^{-3}$<br>$7.8\times10^{-3}$<br>$3.6\times10^{-4}$ | | Wang et al. (2017)<br>Wang et al. (2017)<br>Wang et al. (2017) | Q<br>Q<br>Q | 80, 238<br>80, 239<br>80, 240 |
| MCM:M2PECNO3<br>$C_6H_{13}NO_3$<br>ZLERWGPAWILHHJ-UHFFFAOYSA-N | $7.8\times10^{-3}$<br>$9.6\times10^{-3}$<br>$3.2\times10^{-4}$ | | Wang et al. (2017)<br>Wang et al. (2017)<br>Wang et al. (2017) | Q<br>Q<br>Q | 80, 238<br>80, 239<br>80, 240 |
| MCM:M2PEDNO3<br>$C_6H_{13}NO_3$<br>LFIPHSRTIDHJFT-UHFFFAOYSA-N | $4.6\times10^{-3}$<br>$5.1\times10^{-3}$<br>$6.5\times10^{-4}$ | | Wang et al. (2017)<br>Wang et al. (2017)<br>Wang et al. (2017) | Q<br>Q<br>Q | 80, 238<br>80, 239<br>80, 240 |
| MCM:M33C4NO3<br>$C_6H_{13}NO_3$<br>DENSRMRCUIZNMK-UHFFFAOYSA-N | $4.6\times10^{-3}$<br>$9.8\times10^{-3}$<br>$3.6\times10^{-4}$ | | Wang et al. (2017)<br>Wang et al. (2017)<br>Wang et al. (2017) | Q<br>Q<br>Q | 80, 238<br>80, 239<br>80, 240 |
| MCM:M3PEANO3<br>$C_6H_{13}NO_3$<br>HRVRJIMBNILCGN-UHFFFAOYSA-N | $6.5\times10^{-3}$<br>$1.3\times10^{-2}$<br>$4.7\times10^{-4}$ | | Wang et al. (2017)<br>Wang et al. (2017)<br>Wang et al. (2017) | Q<br>Q<br>Q | 80, 238<br>80, 239<br>80, 240 |
| MCM:M3PEBNO3<br>$C_6H_{13}NO_3$<br>IEHIEWLLOPOHIZ-UHFFFAOYSA-N | $7.8\times10^{-3}$<br>$1.1\times10^{-2}$<br>$4.1\times10^{-4}$ | | Wang et al. (2017)<br>Wang et al. (2017)<br>Wang et al. (2017) | Q<br>Q<br>Q | 80, 238<br>80, 239<br>80, 240 |
| MCM:M3PECNO3<br>$C_6H_{13}NO_3$<br>GFKONLNVJRACGN-UHFFFAOYSA-N | $4.6\times10^{-3}$<br>$5.3\times10^{-3}$<br>$6.2\times10^{-4}$ | | Wang et al. (2017)<br>Wang et al. (2017)<br>Wang et al. (2017) | Q<br>Q<br>Q | 80, 238<br>80, 239<br>80, 240 |
| MCM:PHXN<br>$C_6H_{11}NO_5$<br>ITHMZXMFOKWUTQ-UHFFFAOYSA-N | $2.2$<br>$1.1$<br>$1.8\times10^{-4}$ | | Wang et al. (2017)<br>Wang et al. (2017)<br>Wang et al. (2017) | Q<br>Q<br>Q | 80, 238<br>80, 239<br>80, 240 |
| MCM:HEPTNO3<br>$C_7H_{15}NO_3$<br>RFSNRWONUSBTJN-UHFFFAOYSA-N | $5.4\times10^{-3}$<br>$6.3\times10^{-3}$<br>$3.0\times10^{-4}$ | | Wang et al. (2017)<br>Wang et al. (2017)<br>Wang et al. (2017) | Q<br>Q<br>Q | 80, 238<br>80, 239<br>80, 240 |
| MCM:M2HEXANO3<br>$C_7H_{15}NO_3$<br>YZGBSAOHUZMZSL-UHFFFAOYSA-N | $6.2\times10^{-3}$<br>$5.9\times10^{-3}$<br>$4.0\times10^{-4}$ | | Wang et al. (2017)<br>Wang et al. (2017)<br>Wang et al. (2017) | Q<br>Q<br>Q | 80, 238<br>80, 239<br>80, 240 |
| MCM:M2HEXBNO3<br>$C_7H_{15}NO_3$<br>CORAYZMLWQTCOE-UHFFFAOYSA-N | $3.7\times10^{-3}$<br>$3.9\times10^{-3}$<br>$5.8\times10^{-4}$ | | Wang et al. (2017)<br>Wang et al. (2017)<br>Wang et al. (2017) | Q<br>Q<br>Q | 80, 238<br>80, 239<br>80, 240 |



Table A4.6: Nitrates ($RONO_2$) (...continued)

| Substance Formula (Trivial Name) [CAS Registry Number] InChIKey | $H_s^{cp}$ (at $T^{\ominus}$) $\left[\dfrac{\mathrm{mol}}{\mathrm{m}^3\,\mathrm{Pa}}\right]$ | $\dfrac{\mathrm{d}\ln H_s^{cp}}{\mathrm{d}(1/T)}$ [K] | Reference | Type | Note |
|---|---|---|---|---|---|
| MCM:M3HEXANO3 $C_7H_{15}NO_3$ ZIINVFVZOGZDQJ-UHFFFAOYSA-N | $6.2\times10^{-3}$ $6.6\times10^{-3}$ $3.4\times10^{-4}$ | | Wang et al. (2017) Wang et al. (2017) Wang et al. (2017) | Q Q Q | 80, 238 80, 239 80, 240 |
| MCM:M3HEXBNO3 $C_7H_{15}NO_3$ ADYNXMRWYWNUDX-UHFFFAOYSA-N | $3.7\times10^{-3}$ $4.4\times10^{-3}$ $6.5\times10^{-4}$ | | Wang et al. (2017) Wang et al. (2017) Wang et al. (2017) | Q Q Q | 80, 238 80, 239 80, 240 |
| MCM:PHPTN $C_7H_{13}NO_5$ XCJVDRQLXVBBEC-UHFFFAOYSA-N | $1.8$ $8.9\times10^{-1}$ $2.0\times10^{-4}$ | | Wang et al. (2017) Wang et al. (2017) Wang et al. (2017) | Q Q Q | 80, 238 80, 239 80, 240 |
| MCM:C8BCNO3 $C_8H_{13}NO_3$ OSNOKXIMBITXOJ-UHFFFAOYSA-N | $4.0\times10^{-2}$ $3.2\times10^{-2}$ $3.6\times10^{-3}$ | | Wang et al. (2017) Wang et al. (2017) Wang et al. (2017) | Q Q Q | 80, 238 80, 239 80, 240 |
| MCM:OCTNO3 $C_8H_{17}NO_3$ KDICWCURMCOQTP-UHFFFAOYSA-N | $4.8\times10^{-3}$ $5.0\times10^{-3}$ $2.6\times10^{-4}$ | | Wang et al. (2017) Wang et al. (2017) Wang et al. (2017) | Q Q Q | 80, 238 80, 239 80, 240 |
| MCM:NONNO3 $C_9H_{19}NO_3$ YXUXBXICIGKFLH-UHFFFAOYSA-N | $3.9\times10^{-3}$ $3.8\times10^{-3}$ $2.5\times10^{-4}$ | | Wang et al. (2017) Wang et al. (2017) Wang et al. (2017) | Q Q Q | 80, 238 80, 239 80, 240 |
| MCM:DECNO3 $C_{10}H_{21}NO_3$ VRGDYCVXNCXQKR-UHFFFAOYSA-N | $3.0\times10^{-3}$ $3.0\times10^{-3}$ $2.8\times10^{-4}$ | | Wang et al. (2017) Wang et al. (2017) Wang et al. (2017) | Q Q Q | 80, 238 80, 239 80, 240 |
| MCM:NAPINAOOH $C_{10}H_{17}NO_5$ IKGBGFRUISEOBM-UHFFFAOYSA-N | $1.6\times10^{3}$ $2.3\times10^{2}$ $2.1\times10^{1}$ | | Wang et al. (2017) Wang et al. (2017) Wang et al. (2017) | Q Q Q | 80, 238 80, 239 80, 240 |
| MCM:NAPINBOOH $C_{10}H_{17}NO_5$ RUHGEBIMHPPSGQ-UHFFFAOYSA-N | $1.6\times10^{3}$ $2.5\times10^{2}$ $6.8\times10^{1}$ | | Wang et al. (2017) Wang et al. (2017) Wang et al. (2017) | Q Q Q | 80, 238 80, 239 80, 240 |
| MCM:NBPINAOOH $C_{10}H_{17}NO_5$ VJCKFKSOBAWLFB-UHFFFAOYSA-N | $1.3\times10^{3}$ $4.4\times10^{2}$ $3.5$ | | Wang et al. (2017) Wang et al. (2017) Wang et al. (2017) | Q Q Q | 80, 238 80, 239 80, 240 |
| MCM:NBPINBOOH $C_{10}H_{17}NO_5$ AWECGRRBHMAEGP-UHFFFAOYSA-N | $1.3\times10^{3}$ $3.8\times10^{2}$ $7.8\times10^{1}$ | | Wang et al. (2017) Wang et al. (2017) Wang et al. (2017) | Q Q Q | 80, 238 80, 239 80, 240 |
| MCM:NC91CO3H $C_{10}H_{15}NO_6$ TWWSFGIBZQJYPL-UHFFFAOYSA-N | $1.9\times10^{4}$ $4.8\times10^{2}$ $1.7$ | | Wang et al. (2017) Wang et al. (2017) Wang et al. (2017) | Q Q Q | 80, 238 80, 239 80, 240 |
| MCM:NC91PAN $C_{10}H_{14}N_2O_8$ LCMKIEFNSOTVJY-UHFFFAOYSA-N | $8.0\times10^{2}$ $2.6\times10^{2}$ $9.8\times10^{-4}$ | | Wang et al. (2017) Wang et al. (2017) Wang et al. (2017) | Q Q Q | 80, 238 80, 239 80, 240 |





Table A4.6: Nitrates ($RONO_2$) (. . . continued)

| Substance Formula (Trivial Name) [CAS Registry Number] InChIKey | $H_s^{cp}$ (at $T^\ominus$) $\left[\dfrac{\mathrm{mol}}{\mathrm{m^3\,Pa}}\right]$ | $\dfrac{\mathrm{d\ln} H_s^{cp}}{\mathrm{d}(1/T)}$ [K] | Reference | Type | Note |
|---|---|---|---|---|---|
| MCM:NLIMOOH $C_{10}H_{17}NO_5$ HSXYRSLOXSIYGP-UHFFFAOYSA-N | $1.4\times10^3$ $6.9\times10^2$ $5.3$ | | Wang et al. (2017) Wang et al. (2017) Wang et al. (2017) | Q Q Q | 80, 238 80, 239 80, 240 |
| MCM:UDECNO3 $C_{11}H_{23}NO_3$ KGEKWWIIMVPQKA-UHFFFAOYSA-N | $2.8\times10^{-3}$ $2.4\times10^{-3}$ $2.0\times10^{-4}$ | | Wang et al. (2017) Wang et al. (2017) Wang et al. (2017) | Q Q Q | 80, 238 80, 239 80, 240 |
| MCM:DDECNO3 $C_{12}H_{25}NO_3$ BVAQOKLOUKLOJD-UHFFFAOYSA-N | $2.2\times10^{-3}$ $1.9\times10^{-3}$ $1.9\times10^{-4}$ | | Wang et al. (2017) Wang et al. (2017) Wang et al. (2017) | Q Q Q | 80, 238 80, 239 80, 240 |
| MCM:NBCOOH $C_{15}H_{25}NO_5$ BEFFZATUSVKAHP-UHFFFAOYSA-N | $1.4\times10^3$ $8.7\times10^2$ $3.0\times10^2$ | | Wang et al. (2017) Wang et al. (2017) Wang et al. (2017) | Q Q Q | 80, 238 80, 239 80, 240 |
| MCM:C6H5CH2NO3 $C_7H_7NO_3$ WOIVNLSVAKYSKX-UHFFFAOYSA-N | $4.3\times10^{-1}$ $7.1\times10^{-1}$ $3.6\times10^{-2}$ | | Wang et al. (2017) Wang et al. (2017) Wang et al. (2017) | Q Q Q | 80, 238 80, 239 80, 240 |
| MCM:PBZN $C_7H_5NO_5$ ONDCXZPWEKXYJE-UHFFFAOYSA-N | $1.6\times10^2$ $2.6\times10^1$ $4.4\times10^{-3}$ | | Wang et al. (2017) Wang et al. (2017) Wang et al. (2017) | Q Q Q | 80, 238 80, 239 80, 240 |
| MCM:C6H5C2NO3 $C_8H_9NO_3$ REJUUAZLLYKBCW-UHFFFAOYSA-N | $3.6\times10^{-1}$ $5.6\times10^{-1}$ $4.2\times10^{-2}$ | | Wang et al. (2017) Wang et al. (2017) Wang et al. (2017) | Q Q Q | 80, 238 80, 239 80, 240 |
| MCM:C6H5CH2PAN $C_8H_7NO_5$ KKRIGRPRRNIVGL-UHFFFAOYSA-N | $1.4\times10^2$ $4.7\times10^1$ $1.2\times10^{-2}$ | | Wang et al. (2017) Wang et al. (2017) Wang et al. (2017) | Q Q Q | 80, 238 80, 239 80, 240 |
| MCM:MXYLNO3 $C_8H_9NO_3$ OIEIRDPNAPXLBC-UHFFFAOYSA-N | $2.5\times10^{-1}$ $6.5\times10^{-1}$ $3.1\times10^{-2}$ | | Wang et al. (2017) Wang et al. (2017) Wang et al. (2017) | Q Q Q | 80, 238 80, 239 80, 240 |
| MCM:MXYLPAN $C_8H_7NO_5$ HJGURQOGPAEQPE-UHFFFAOYSA-N | $8.9\times10^1$ $2.3\times10^1$ $4.1\times10^{-3}$ | | Wang et al. (2017) Wang et al. (2017) Wang et al. (2017) | Q Q Q | 80, 238 80, 239 80, 240 |
| MCM:NSTYRENOOH $C_8H_9NO_5$ LKBHTMHNYICJQS-UHFFFAOYSA-N | $3.0\times10^4$ $1.1\times10^4$ $1.1\times10^2$ | | Wang et al. (2017) Wang et al. (2017) Wang et al. (2017) | Q Q Q | 80, 238 80, 239 80, 240 |
| MCM:OXYLNO3 $C_8H_9NO_3$ WKRHODAGVPDBTG-UHFFFAOYSA-N | $2.5\times10^{-1}$ $7.1\times10^{-1}$ $2.6\times10^{-2}$ | | Wang et al. (2017) Wang et al. (2017) Wang et al. (2017) | Q Q Q | 80, 238 80, 239 80, 240 |
| MCM:OXYLPAN $C_8H_7NO_5$ TXXXSKKGTSQWKQ-UHFFFAOYSA-N | $8.9\times10^1$ $3.1\times10^1$ $3.2\times10^{-3}$ | | Wang et al. (2017) Wang et al. (2017) Wang et al. (2017) | Q Q Q | 80, 238 80, 239 80, 240 |



Table A4.6: Nitrates ($RONO_2$) (. . . continued)

| Substance<br>Formula<br>(Trivial Name)<br>[CAS Registry Number]<br>InChIKey | $H_s^{cp}$<br>(at $T^{\ominus}$)<br>$\left[\dfrac{\text{mol}}{\text{m}^3\,\text{Pa}}\right]$ | $\dfrac{\text{d}\ln H_s^{cp}}{\text{d}(1/T)}$<br><br>[K] | Reference | Type | Note |
|---|---|---|---|---|---|
| MCM:PXYLNO3<br>$C_8H_9NO_3$<br>SHZSDLPSACZYMH-UHFFFAOYSA-N | $2.5\times10^{-1}$<br>$7.8\times10^{-1}$<br>$3.6\times10^{-2}$ | | Wang et al. (2017)<br>Wang et al. (2017)<br>Wang et al. (2017) | Q<br>Q<br>Q | 80, 238<br>80, 239<br>80, 240 |
| MCM:PXYLPAN<br>$C_8H_7NO_5$<br>ZGFUWGCXWYZODV-UHFFFAOYSA-N | $8.9\times10^{1}$<br>$2.6\times10^{1}$<br>$6.5\times10^{-3}$ | | Wang et al. (2017)<br>Wang et al. (2017)<br>Wang et al. (2017) | Q<br>Q<br>Q | 80, 238<br>80, 239<br>80, 240 |
| MCM:ETOLNO3<br>$C_9H_{11}NO_3$<br>LYNFROWVHACAMO-UHFFFAOYSA-N | $2.3\times10^{-1}$<br>$3.2\times10^{-1}$<br>$1.9\times10^{-2}$ | | Wang et al. (2017)<br>Wang et al. (2017)<br>Wang et al. (2017) | Q<br>Q<br>Q | 80, 238<br>80, 239<br>80, 240 |
| MCM:PHC3NO3<br>$C_9H_{11}NO_3$<br>IFDAHDBLKMQHMM-UHFFFAOYSA-N | $3.3\times10^{-1}$<br>$2.2\times10^{-1}$<br>$1.8\times10^{-2}$ | | Wang et al. (2017)<br>Wang et al. (2017)<br>Wang et al. (2017) | Q<br>Q<br>Q | 80, 238<br>80, 239<br>80, 240 |
| MCM:PHIC3NO3<br>$C_9H_{11}NO_3$<br>YCCLKZYFPBVILU-UHFFFAOYSA-N | $2.2\times10^{-1}$<br>$1.5\times10^{-1}$<br>$3.8\times10^{-2}$ | | Wang et al. (2017)<br>Wang et al. (2017)<br>Wang et al. (2017) | Q<br>Q<br>Q | 80, 238<br>80, 239<br>80, 240 |
| MCM:TM123BNO3<br>$C_9H_{11}NO_3$<br>GDXSVUXDQKEMHM-UHFFFAOYSA-N | $1.6\times10^{-1}$<br>$9.8\times10^{-1}$<br>$3.0\times10^{-2}$ | | Wang et al. (2017)<br>Wang et al. (2017)<br>Wang et al. (2017) | Q<br>Q<br>Q | 80, 238<br>80, 239<br>80, 240 |
| MCM:TM123BPAN<br>$C_9H_9NO_5$<br>VMALQUVLPIFIEM-UHFFFAOYSA-N | $5.9\times10^{1}$<br>$3.9\times10^{1}$<br>$4.5\times10^{-3}$ | | Wang et al. (2017)<br>Wang et al. (2017)<br>Wang et al. (2017) | Q<br>Q<br>Q | 80, 238<br>80, 239<br>80, 240 |
| MCM:TM124BNO3<br>$C_9H_{11}NO_3$<br>VLXWVVSZKXVGNM-UHFFFAOYSA-N | $1.6\times10^{-1}$<br>$9.3\times10^{-1}$<br>$3.6\times10^{-2}$ | | Wang et al. (2017)<br>Wang et al. (2017)<br>Wang et al. (2017) | Q<br>Q<br>Q | 80, 238<br>80, 239<br>80, 240 |
| MCM:TM124BPAN<br>$C_9H_9NO_5$<br>SSMPYZHCWJGIJH-UHFFFAOYSA-N | $5.9\times10^{1}$<br>$3.0\times10^{1}$<br>$7.4\times10^{-3}$ | | Wang et al. (2017)<br>Wang et al. (2017)<br>Wang et al. (2017) | Q<br>Q<br>Q | 80, 238<br>80, 239<br>80, 240 |
| MCM:TMBNO3<br>$C_9H_{11}NO_3$<br>CHDZRXQPKVNDRJ-UHFFFAOYSA-N | $1.6\times10^{-1}$<br>$5.5\times10^{-1}$<br>$2.8\times10^{-2}$ | | Wang et al. (2017)<br>Wang et al. (2017)<br>Wang et al. (2017) | Q<br>Q<br>Q | 80, 238<br>80, 239<br>80, 240 |
| MCM:TMBPAN<br>$C_9H_9NO_5$<br>QURBGWBLEPWFEF-UHFFFAOYSA-N | $5.9\times10^{1}$<br>$1.9\times10^{1}$<br>$4.0\times10^{-3}$ | | Wang et al. (2017)<br>Wang et al. (2017)<br>Wang et al. (2017) | Q<br>Q<br>Q | 80, 238<br>80, 239<br>80, 240 |
| MCM:DM35EBNO3<br>$C_{10}H_{13}NO_3$<br>CZPIEIXHJQQIGZ-UHFFFAOYSA-N | $1.4\times10^{-1}$<br>$2.4\times10^{-1}$<br>$1.7\times10^{-2}$ | | Wang et al. (2017)<br>Wang et al. (2017)<br>Wang et al. (2017) | Q<br>Q<br>Q | 80, 238<br>80, 239<br>80, 240 |
| MCM:EMPHPAN<br>$C_{10}H_{11}NO_5$<br>DPRDWMVUHNTRQI-UHFFFAOYSA-N | $4.9\times10^{1}$<br>$1.1\times10^{1}$<br>$3.2\times10^{-3}$ | | Wang et al. (2017)<br>Wang et al. (2017)<br>Wang et al. (2017) | Q<br>Q<br>Q | 80, 238<br>80, 239<br>80, 240 |



Table A4.6: Nitrates ($RONO_2$) (...continued)

| Substance Formula (Trivial Name) [CAS Registry Number] InChIKey | $H_s^{cp}$ (at $T^\ominus$) $\left[\dfrac{\text{mol}}{\text{m}^3\,\text{Pa}}\right]$ | $\dfrac{\text{d}\ln H_s^{cp}}{\text{d}(1/T)}$ [K] | Reference | Type | Note |
|---|---|---|---|---|---|
| MCM:DE35TNO3 | $1.1\times10^{-1}$ | | Wang et al. (2017) | Q | 80, 238 |
| $C_{11}H_{15}NO_3$ | $1.5\times10^{-1}$ | | Wang et al. (2017) | Q | 80, 239 |
| XLLLSPDIGUQFTC-UHFFFAOYSA-N | $1.6\times10^{-2}$ | | Wang et al. (2017) | Q | 80, 240 |
| MCM:PHAN | $6.2\times10^{2}$ | | Wang et al. (2017) | Q | 80, 238 |
| $C_2H_3NO_6$ | $9.6\times10^{3}$ | | Wang et al. (2017) | Q | 80, 239 |
| KMDWHTZYVJXWAH-UHFFFAOYSA-N | $6.6\times10^{-1}$ | | Wang et al. (2017) | Q | 80, 240 |
| MCM:A2PAN | $7.3\times10^{5}$ | | Wang et al. (2017) | Q | 80, 238 |
| $C_3H_5NO_7$ | $2.3\times10^{6}$ | | Wang et al. (2017) | Q | 80, 239 |
| RQFAVCUBLOXLKI-UHFFFAOYSA-N | $2.2\times10^{1}$ | | Wang et al. (2017) | Q | 80, 240 |
| MCM:C3PAN1 | $1.1\times10^{4}$ | | Wang et al. (2017) | Q | 80, 238 |
| $C_3H_5NO_6$ | $3.2\times10^{4}$ | | Wang et al. (2017) | Q | 80, 239 |
| ZDHDBTZUUXEIQG-UHFFFAOYSA-N | $6.6\times10^{-1}$ | | Wang et al. (2017) | Q | 80, 240 |
| MCM:HO1C3NO3 | $2.8\times10^{1}$ | | Wang et al. (2017) | Q | 80, 238 |
| $C_3H_7NO_4$ | $1.2\times10^{3}$ | | Wang et al. (2017) | Q | 80, 239 |
| [100502-66-7] | $6.3$ | | Wang et al. (2017) | Q | 80, 240 |
| PTMLFFXFTRSBJW-UHFFFAOYSA-N | $2.0\times10^{1}$ | | Raventos-Duran et al. (2010) | Q | 271, 243 |
| | $3.9\times10^{2}$ | | Raventos-Duran et al. (2010) | Q | 244 |
| | $7.8\times10^{2}$ | | Raventos-Duran et al. (2010) | Q | 245 |
| MCM:IPROPOLPAN | $5.9\times10^{2}$ | | Wang et al. (2017) | Q | 80, 238 |
| $C_3H_5NO_6$ | $1.1\times10^{4}$ | | Wang et al. (2017) | Q | 80, 239 |
| VGQGEUUGHUVAJJ-UHFFFAOYSA-N | $2.0\times10^{-1}$ | | Wang et al. (2017) | Q | 80, 240 |
| MCM:BUTDANO3 | $1.0\times10^{2}$ | | Wang et al. (2017) | Q | 80, 238 |
| $C_4H_7NO_4$ | $2.3\times10^{3}$ | | Wang et al. (2017) | Q | 80, 239 |
| JVISETCEJQRRQM-UHFFFAOYSA-N | $3.0\times10^{1}$ | | Wang et al. (2017) | Q | 80, 240 |
| MCM:BUTDBNO3 | $9.8\times10^{1}$ | | Wang et al. (2017) | Q | 80, 238 |
| $C_4H_7NO_4$ | $2.5\times10^{2}$ | | Wang et al. (2017) | Q | 80, 239 |
| MVWTUBMDIKBQOB-UHFFFAOYSA-N | $3.0$ | | Wang et al. (2017) | Q | 80, 240 |
| MCM:C4PAN1 | $8.7\times10^{3}$ | | Wang et al. (2017) | Q | 80, 238 |
| $C_4H_7NO_6$ | $5.5\times10^{4}$ | | Wang et al. (2017) | Q | 80, 239 |
| ZAYFYZJEFHQRME-UHFFFAOYSA-N | $2.1$ | | Wang et al. (2017) | Q | 80, 240 |
| MCM:C4PAN2 | $1.3\times10^{6}$ | | Wang et al. (2017) | Q | 80, 238 |
| $C_4H_7NO_7$ | $1.7\times10^{7}$ | | Wang et al. (2017) | Q | 80, 239 |
| KFPIYXZEKUAVMP-UHFFFAOYSA-N | $1.2\times10^{2}$ | | Wang et al. (2017) | Q | 80, 240 |
| MCM:C4PAN3 | $1.0\times10^{4}$ | | Wang et al. (2017) | Q | 80, 238 |
| $C_4H_7NO_6$ | $3.6\times10^{4}$ | | Wang et al. (2017) | Q | 80, 239 |
| PIBQEVCOTXTKAE-UHFFFAOYSA-N | $4.2\times10^{-1}$ | | Wang et al. (2017) | Q | 80, 240 |
| MCM:C4PAN4 | $1.0\times10^{4}$ | | Wang et al. (2017) | Q | 80, 238 |
| $C_4H_7NO_6$ | $2.0\times10^{4}$ | | Wang et al. (2017) | Q | 80, 239 |
| NXHSYZWHYBOEBH-UHFFFAOYSA-N | $4.0\times10^{-1}$ | | Wang et al. (2017) | Q | 80, 240 |



Table A4.6: Nitrates ($RONO_2$) (... continued)

| Substance<br>Formula<br>(Trivial Name)<br>[CAS Registry Number]<br>InChIKey | $H_s^{cp}$<br>(at $T^\ominus$)<br>$\left[\dfrac{\mathrm{mol}}{\mathrm{m^3\,Pa}}\right]$ | $\dfrac{\mathrm{d}\ln H_s^{cp}}{\mathrm{d}(1/T)}$<br><br>[K] | Reference | Type | Note |
|---|---|---|---|---|---|
| MCM:C4PAN5 | $3.2\times10^2$ | | Wang et al. (2017) | Q | 80, 238 |
| $C_4H_7NO_6$ | $4.1\times10^3$ | | Wang et al. (2017) | Q | 80, 239 |
| BZGCEXLIVXRKKX-UHFFFAOYSA-N | $5.5\times10^{-2}$ | | Wang et al. (2017) | Q | 80, 240 |
| MCM:C4PAN7 | $3.6\times10^4$ | | Wang et al. (2017) | Q | 80, 238 |
| $C_4H_5NO_6$ | $1.2\times10^5$ | | Wang et al. (2017) | Q | 80, 239 |
| YKQOPDHVAHDODD-UHFFFAOYSA-N | $2.5\times10^1$ | | Wang et al. (2017) | Q | 80, 240 |
| MCM:C4PAN8 | $1.5\times10^3$ | | Wang et al. (2017) | Q | 80, 238 |
| $C_4H_5NO_6$ | $7.1\times10^3$ | | Wang et al. (2017) | Q | 80, 239 |
| QZCDOTFCQHJUGH-UHFFFAOYSA-N | $6.5\times10^{-2}$ | | Wang et al. (2017) | Q | 80, 240 |
| MCM:C56NO3 | $8.0\times10^4$ | | Wang et al. (2017) | Q | 80, 238 |
| $C_4H_9NO_5$ | $9.1\times10^5$ | | Wang et al. (2017) | Q | 80, 239 |
| XYUWBXDQANMXIL-UHFFFAOYSA-N | $1.7\times10^2$ | | Wang et al. (2017) | Q | 80, 240 |
| MCM:HMPAN | $2.1\times10^4$ | | Wang et al. (2017) | Q | 80, 238 |
| $C_4H_5NO_6$ | $1.4\times10^4$ | | Wang et al. (2017) | Q | 80, 239 |
| XPXMBKPWAOWOBP-UHFFFAOYSA-N | $6.8\times10^{-1}$ | | Wang et al. (2017) | Q | 80, 240 |
| MCM:HO13C4NO3 | $8.9\times10^4$ | | Wang et al. (2017) | Q | 80, 238 |
| $C_4H_9NO_5$ | $1.0\times10^6$ | | Wang et al. (2017) | Q | 80, 239 |
| IOYLZWMTXYBAPI-UHFFFAOYSA-N | $3.2\times10^3$ | | Wang et al. (2017) | Q | 80, 240 |
| MCM:HO3C3PAN | $5.1\times10^2$ | | Wang et al. (2017) | Q | 80, 238 |
| $C_4H_7NO_6$ | $6.2\times10^3$ | | Wang et al. (2017) | Q | 80, 239 |
| JGMDPJZYJFPSOE-UHFFFAOYSA-N | $9.1\times10^{-2}$ | | Wang et al. (2017) | Q | 80, 240 |
| MCM:IBUTOLBNO3 | $2.2\times10^1$ | | Wang et al. (2017) | Q | 80, 238 |
| $C_4H_9NO_4$ | $1.0\times10^2$ | | Wang et al. (2017) | Q | 80, 239 |
| ILMPSGIQBJFTRG-UHFFFAOYSA-N | $6.8\times10^{-1}$ | | Wang et al. (2017) | Q | 80, 240 |
| MCM:IBUTOLCNO3 | $2.6\times10^1$ | | Wang et al. (2017) | Q | 80, 238 |
| $C_4H_9NO_4$ | $6.9\times10^2$ | | Wang et al. (2017) | Q | 80, 239 |
| OARBGYKXUANYLR-UHFFFAOYSA-N | $3.5$ | | Wang et al. (2017) | Q | 80, 240 |
| MCM:MACRNBPAN | $5.0\times10^4$ | | Wang et al. (2017) | Q | 80, 238 |
| $C_4H_6N_2O_9$ | $7.6\times10^5$ | | Wang et al. (2017) | Q | 80, 239 |
| HHPQSZXZUAZGAQ-UHFFFAOYSA-N | $1.8\times10^{-1}$ | | Wang et al. (2017) | Q | 80, 240 |
| MCM:MACRNCO3H | $2.4\times10^7$ | | Wang et al. (2017) | Q | 80, 238 |
| $C_4H_7NO_7$ | $2.3\times10^6$ | | Wang et al. (2017) | Q | 80, 239 |
| VCAMMMQHKHRNFT-UHFFFAOYSA-N | $1.7\times10^1$ | | Wang et al. (2017) | Q | 80, 240 |
| MCM:MACRNPAN | $9.8\times10^5$ | | Wang et al. (2017) | Q | 80, 238 |
| $C_4H_6N_2O_9$ | $2.8\times10^5$ | | Wang et al. (2017) | Q | 80, 239 |
| YVNHVFURNFQJQM-UHFFFAOYSA-N | $1.2\times10^{-2}$ | | Wang et al. (2017) | Q | 80, 240 |
| MCM:NBUTDAOH | $9.8\times10^1$ | | Wang et al. (2017) | Q | 80, 238 |
| $C_4H_7NO_4$ | $2.9\times10^2$ | | Wang et al. (2017) | Q | 80, 239 |
| ZANUSWCYRLKDAN-UHFFFAOYSA-N | $2.0$ | | Wang et al. (2017) | Q | 80, 240 |





Table A4.6: Nitrates ($RONO_2$) (...continued)

| Substance Formula (Trivial Name) [CAS Registry Number] InChIKey | $H_s^{cp}$ (at $T^\ominus$) $\left[\dfrac{\text{mol}}{\text{m}^3\,\text{Pa}}\right]$ | $\dfrac{\text{d}\ln H_s^{cp}}{\text{d}(1/T)}$ [K] | Reference | Type | Note |
|---|---|---|---|---|---|
| MCM:TBUTOLNO3 | $2.2\times10^1$ | | Wang et al. (2017) | Q | 80, 238 |
| $C_4H_9NO_4$ | $1.4\times10^2$ | | Wang et al. (2017) | Q | 80, 239 |
| SPXXYWSDFUWLEP-UHFFFAOYSA-N | $9.8\times10^{-1}$ | | Wang et al. (2017) | Q | 80, 240 |
| MCM:C3M3OH2PAN | $4.8\times10^2$ | | Wang et al. (2017) | Q | 80, 238 |
| $C_5H_9NO_6$ | $4.9\times10^3$ | | Wang et al. (2017) | Q | 80, 239 |
| NSGPYXAFGPXNMV-UHFFFAOYSA-N | $2.8\times10^{-2}$ | | Wang et al. (2017) | Q | 80, 240 |
| MCM:C46PAN | $1.6\times10^4$ | | Wang et al. (2017) | Q | 80, 238 |
| $C_5H_7NO_6$ | $7.4\times10^4$ | | Wang et al. (2017) | Q | 80, 239 |
| WDMYRNZGJSFESH-UHFFFAOYSA-N | $2.0$ | | Wang et al. (2017) | Q | 80, 240 |
| MCM:C4OH2CPAN | $3.7\times10^5$ | | Wang et al. (2017) | Q | 80, 238 |
| $C_5H_9NO_7$ | $3.5\times10^6$ | | Wang et al. (2017) | Q | 80, 239 |
| INCMEOVDRBCULR-UHFFFAOYSA-N | $1.1$ | | Wang et al. (2017) | Q | 80, 240 |
| MCM:C4OHPAN | $4.3\times10^2$ | | Wang et al. (2017) | Q | 80, 238 |
| $C_5H_9NO_6$ | $3.7\times10^3$ | | Wang et al. (2017) | Q | 80, 239 |
| GAAOIPAQIRGTBI-UHFFFAOYSA-N | $5.3\times10^{-2}$ | | Wang et al. (2017) | Q | 80, 240 |
| MCM:C524NO3 | $1.7\times10^5$ | | Wang et al. (2017) | Q | 80, 238 |
| $C_5H_9NO_5$ | $2.6\times10^6$ | | Wang et al. (2017) | Q | 80, 239 |
| IRMGDWDXIVCKPE-UHFFFAOYSA-N | $2.2\times10^3$ | | Wang et al. (2017) | Q | 80, 240 |
| MCM:C52NO3 | $2.0\times10^1$ | | Wang et al. (2017) | Q | 80, 238 |
| $C_5H_{11}NO_4$ | $5.0\times10^2$ | | Wang et al. (2017) | Q | 80, 239 |
| NBCYUKLMSBOGIC-UHFFFAOYSA-N | $6.5$ | | Wang et al. (2017) | Q | 80, 240 |
| MCM:C54NO3 | $4.9\times10^4$ | | Wang et al. (2017) | Q | 80, 238 |
| $C_5H_{11}NO_5$ | $3.8\times10^5$ | | Wang et al. (2017) | Q | 80, 239 |
| OXSZOTYJVSIXIT-UHFFFAOYSA-N | $9.1\times10^2$ | | Wang et al. (2017) | Q | 80, 240 |
| MCM:C57NO3CO3H | $2.4\times10^{10}$ | | Wang et al. (2017) | Q | 80, 238 |
| $C_5H_9NO_8$ | $1.4\times10^9$ | | Wang et al. (2017) | Q | 80, 239 |
| BPRAPKBGVRGZEK-UHFFFAOYSA-N | $3.6\times10^3$ | | Wang et al. (2017) | Q | 80, 240 |
| MCM:C57NO3PAN | $9.8\times10^8$ | | Wang et al. (2017) | Q | 80, 238 |
| $C_5H_8N_2O_{10}$ | $2.7\times10^8$ | | Wang et al. (2017) | Q | 80, 239 |
| QHZGCLHPLOENFL-UHFFFAOYSA-N | $2.5\times10^{-1}$ | | Wang et al. (2017) | Q | 80, 240 |
| MCM:C58NO3CO3H | $3.5\times10^9$ | | Wang et al. (2017) | Q | 80, 238 |
| $C_5H_9NO_8$ | $5.8\times10^7$ | | Wang et al. (2017) | Q | 80, 239 |
| NFQMHUFQLLKQQS-UHFFFAOYSA-N | $5.1\times10^1$ | | Wang et al. (2017) | Q | 80, 240 |
| MCM:C58NO3PAN | $1.4\times10^8$ | | Wang et al. (2017) | Q | 80, 238 |
| $C_5H_8N_2O_{10}$ | $4.2\times10^8$ | | Wang et al. (2017) | Q | 80, 239 |
| VIMTZSSJOGPGII-UHFFFAOYSA-N | $8.5\times10^{-1}$ | | Wang et al. (2017) | Q | 80, 240 |
| MCM:C5PAN10 | $1.2\times10^6$ | | Wang et al. (2017) | Q | 80, 238 |
| $C_5H_9NO_7$ | $2.5\times10^7$ | | Wang et al. (2017) | Q | 80, 239 |
| YRCMXGZRWLAVBH-UHFFFAOYSA-N | $9.6\times10^1$ | | Wang et al. (2017) | Q | 80, 240 |



Table A4.6: Nitrates ($RONO_2$) (...continued)

| Substance Formula (Trivial Name) [CAS Registry Number] InChIKey | $H_s^{cp}$ (at $T^{\ominus}$) $\left[\dfrac{\mathrm{mol}}{\mathrm{m^3\,Pa}}\right]$ | $\dfrac{\mathrm{d}\ln H_s^{cp}}{\mathrm{d}(1/T)}$ [K] | Reference | Type | Note |
|---|---|---|---|---|---|
| MCM:C5PAN11 $C_5H_9NO_6$ ISSHIIFFUOEHLT-UHFFFAOYSA-N | $5.8\times10^3$ $2.4\times10^4$ $1.4\times10^{-1}$ | | Wang et al. (2017) Wang et al. (2017) Wang et al. (2017) | Q Q Q | 80, 238 80, 239 80, 240 |
| MCM:C5PAN12 $C_5H_9NO_7$ VRGURBHINCFLEI-UHFFFAOYSA-N | $7.3\times10^5$ $1.0\times10^7$ $9.3\times10^1$ | | Wang et al. (2017) Wang et al. (2017) Wang et al. (2017) | Q Q Q | 80, 238 80, 239 80, 240 |
| MCM:C5PAN13 $C_5H_9NO_7$ GDYHQRWKNLRCBR-UHFFFAOYSA-N | $1.2\times10^6$ $2.0\times10^7$ $5.8\times10^1$ | | Wang et al. (2017) Wang et al. (2017) Wang et al. (2017) | Q Q Q | 80, 238 80, 239 80, 240 |
| MCM:C5PAN14 $C_5H_9NO_6$ KAYGTWBSLGDXRV-UHFFFAOYSA-N | $9.3\times10^3$ $2.2\times10^4$ $5.8\times10^{-2}$ | | Wang et al. (2017) Wang et al. (2017) Wang et al. (2017) | Q Q Q | 80, 238 80, 239 80, 240 |
| MCM:C5PAN15 $C_5H_9NO_6$ DDVUSZKFLFMJLG-UHFFFAOYSA-N | $5.8\times10^3$ $9.3\times10^3$ $5.0\times10^{-2}$ | | Wang et al. (2017) Wang et al. (2017) Wang et al. (2017) | Q Q Q | 80, 238 80, 239 80, 240 |
| MCM:C5PAN17 $C_5H_7NO_6$ WOHMTHRFBCBPEN-UHFFFAOYSA-N | $2.3\times10^4$ $1.3\times10^5$ $2.0\times10^1$ | | Wang et al. (2017) Wang et al. (2017) Wang et al. (2017) | Q Q Q | 80, 238 80, 239 80, 240 |
| MCM:C5PAN19 $C_5H_7NO_6$ SXXBKHKGTQGHOF-UHFFFAOYSA-N | $2.3\times10^4$ $1.0\times10^5$ $1.3\times10^1$ | | Wang et al. (2017) Wang et al. (2017) Wang et al. (2017) | Q Q Q | 80, 238 80, 239 80, 240 |
| MCM:C5PAN1 $C_5H_9NO_6$ BXAPPDLQCWAVAO-UHFFFAOYSA-N | $8.3\times10^3$ $4.0\times10^4$ $1.2$ | | Wang et al. (2017) Wang et al. (2017) Wang et al. (2017) | Q Q Q | 80, 238 80, 239 80, 240 |
| MCM:C5PAN3 $C_5H_9NO_6$ GANOVMQBFKDCJS-UHFFFAOYSA-N | $8.3\times10^3$ $3.3\times10^4$ $1.3$ | | Wang et al. (2017) Wang et al. (2017) Wang et al. (2017) | Q Q Q | 80, 238 80, 239 80, 240 |
| MCM:C5PAN5 $C_5H_9NO_6$ SNADUXVVPGYKTN-UHFFFAOYSA-N | $8.3\times10^3$ $3.0\times10^4$ $2.2$ | | Wang et al. (2017) Wang et al. (2017) Wang et al. (2017) | Q Q Q | 80, 238 80, 239 80, 240 |
| MCM:C5PAN8 $C_5H_9NO_6$ AVADEHRLOLZNNT-UHFFFAOYSA-N | $8.3\times10^3$ $1.9\times10^4$ $7.6\times10^{-2}$ | | Wang et al. (2017) Wang et al. (2017) Wang et al. (2017) | Q Q Q | 80, 238 80, 239 80, 240 |
| MCM:H2M3C4NO3 $C_5H_{11}NO_4$ KRCHLCXLZXFPMO-UHFFFAOYSA-N | $2.5\times10^1$ $5.0\times10^2$ $1.5$ | | Wang et al. (2017) Wang et al. (2017) Wang et al. (2017) | Q Q Q | 80, 238 80, 239 80, 240 |
| MCM:HM22C3NO3 $C_5H_{11}NO_4$ GJZIQWDDCYOUOR-UHFFFAOYSA-N | $1.4\times10^1$ $3.2\times10^2$ $1.1$ | | Wang et al. (2017) Wang et al. (2017) Wang et al. (2017) | Q Q Q | 80, 238 80, 239 80, 240 |





Table A4.6: Nitrates ($RONO_2$) (...continued)

| Substance / Formula / (Trivial Name) / [CAS Registry Number] / InChIKey | $H_s^{cp}$ (at $T^{\ominus}$) $\left[\dfrac{\mathrm{mol}}{\mathrm{m^3\,Pa}}\right]$ | $\dfrac{\mathrm{d}\ln H_s^{cp}}{\mathrm{d}(1/T)}$ [K] | Reference | Type | Note |
|---|---|---|---|---|---|
| MCM:HM2C43NO3 | $2.5\times10^1$ | | Wang et al. (2017) | Q | 80, 238 |
| $C_5H_{11}NO_4$ | $4.1\times10^2$ | | Wang et al. (2017) | Q | 80, 239 |
| OLLXHFMQLUGEEM-UHFFFAOYSA-N | 2.5 | | Wang et al. (2017) | Q | 80, 240 |
| MCM:HM33C3NO3 | $1.4\times10^1$ | | Wang et al. (2017) | Q | 80, 238 |
| $C_5H_{11}NO_4$ | $2.0\times10^2$ | | Wang et al. (2017) | Q | 80, 239 |
| LCUGTHWCLWAZIN-UHFFFAOYSA-N | 5.3 | | Wang et al. (2017) | Q | 80, 240 |
| MCM:HO13C5NO3 | $8.3\times10^4$ | | Wang et al. (2017) | Q | 80, 238 |
| $C_5H_{11}NO_5$ | $5.4\times10^5$ | | Wang et al. (2017) | Q | 80, 239 |
| HRXGEJHIRPMYKD-UHFFFAOYSA-N | $1.6\times10^3$ | | Wang et al. (2017) | Q | 80, 240 |
| MCM:HO24C5NO3 | $8.3\times10^4$ | | Wang et al. (2017) | Q | 80, 238 |
| $C_5H_{11}NO_5$ | $5.8\times10^5$ | | Wang et al. (2017) | Q | 80, 239 |
| GDKLMXJUXHUPHP-UHFFFAOYSA-N | $1.4\times10^3$ | | Wang et al. (2017) | Q | 80, 240 |
| MCM:HO2C54NO3 | $2.5\times10^1$ | | Wang et al. (2017) | Q | 80, 238 |
| $C_5H_{11}NO_4$ | $2.8\times10^2$ | | Wang et al. (2017) | Q | 80, 239 |
| BDOVYOMPHMUDCA-UHFFFAOYSA-N | 1.8 | | Wang et al. (2017) | Q | 80, 240 |
| MCM:HO2M2C4NO3 | $1.4\times10^1$ | | Wang et al. (2017) | Q | 80, 238 |
| $C_5H_{11}NO_4$ | $3.6\times10^2$ | | Wang et al. (2017) | Q | 80, 239 |
| JLEBTGVTEJACMC-UHFFFAOYSA-N | 1.6 | | Wang et al. (2017) | Q | 80, 240 |
| MCM:HO3C5NO3 | $2.0\times10^1$ | | Wang et al. (2017) | Q | 80, 238 |
| $C_5H_{11}NO_4$ | $3.8\times10^2$ | | Wang et al. (2017) | Q | 80, 239 |
| VSJMKXXNGJNYTF-UHFFFAOYSA-N | 1.4 | | Wang et al. (2017) | Q | 80, 240 |
| MCM:INAHCO3H | $1.2\times10^9$ | | Wang et al. (2017) | Q | 80, 238 |
| $C_5H_9NO_8$ | $4.9\times10^7$ | | Wang et al. (2017) | Q | 80, 239 |
| NANYPKIUOGNJFQ-UHFFFAOYSA-N | $5.8\times10^2$ | | Wang et al. (2017) | Q | 80, 240 |
| MCM:INAHPAN | $5.0\times10^7$ | | Wang et al. (2017) | Q | 80, 238 |
| $C_5H_8N_2O_{10}$ | $6.8\times10^8$ | | Wang et al. (2017) | Q | 80, 239 |
| ISRNXRSYOJKFEQ-UHFFFAOYSA-N | 5.0 | | Wang et al. (2017) | Q | 80, 240 |
| MCM:INAHPCO3H | $1.8\times10^{12}$ | | Wang et al. (2017) | Q | 80, 238 |
| $C_5H_9NO_9$ | $2.5\times10^9$ | | Wang et al. (2017) | Q | 80, 239 |
| ZYWYMWCOZFANRV-UHFFFAOYSA-N | $3.8\times10^2$ | | Wang et al. (2017) | Q | 80, 240 |
| MCM:INAHPPAN | $8.0\times10^{10}$ | | Wang et al. (2017) | Q | 80, 238 |
| $C_5H_8N_2O_{11}$ | $1.1\times10^9$ | | Wang et al. (2017) | Q | 80, 239 |
| ZXVKOORKVQYPPA-UHFFFAOYSA-N | 3.4 | | Wang et al. (2017) | Q | 80, 240 |
| MCM:INANCO3H | $2.7\times10^9$ | | Wang et al. (2017) | Q | 80, 238 |
| $C_5H_8N_2O_{10}$ | $2.6\times10^8$ | | Wang et al. (2017) | Q | 80, 239 |
| HKZKRKUNOXITIG-UHFFFAOYSA-N | $3.1\times10^1$ | | Wang et al. (2017) | Q | 80, 240 |
| MCM:INANO3 | $7.8\times10^6$ | | Wang et al. (2017) | Q | 80, 238 |
| $C_5H_{10}N_2O_8$ | $2.6\times10^7$ | | Wang et al. (2017) | Q | 80, 239 |
| VVVQXVVEBQASSQ-UHFFFAOYSA-N | $1.7\times10^2$ | | Wang et al. (2017) | Q | 80, 240 |



Table A4.6: Nitrates ($RONO_2$) (...continued)

| Substance<br>Formula<br>(Trivial Name)<br>[CAS Registry Number]<br>InChIKey | $H_s^{cp}$<br>(at $T^\ominus$)<br>$\left[\dfrac{\mathrm{mol}}{\mathrm{m^3\,Pa}}\right]$ | $\dfrac{\mathrm{d}\ln H_s^{cp}}{\mathrm{d}(1/T)}$<br><br>[K] | Reference | Type | Note |
|---|---|---|---|---|---|
| MCM:INANPAN | $1.2\times10^8$ | | Wang et al. (2017) | Q | 80, 238 |
| $C_5H_7N_3O_{12}$ | $3.7\times10^7$ | | Wang et al. (2017) | Q | 80, 239 |
| PTXQCTHWIXKTGJ-UHFFFAOYSA-N | $4.7\times10^{-2}$ | | Wang et al. (2017) | Q | 80, 240 |
| MCM:INAOH | $4.0\times10^7$ | | Wang et al. (2017) | Q | 80, 238 |
| $C_5H_{11}NO_6$ | $2.9\times10^8$ | | Wang et al. (2017) | Q | 80, 239 |
| KOIOFJRFFFLZDV-UHFFFAOYSA-N | $3.2\times10^3$ | | Wang et al. (2017) | Q | 80, 240 |
| MCM:INAOOH | $5.3\times10^9$ | | Wang et al. (2017) | Q | 80, 238 |
| $C_5H_{11}NO_7$ | $4.7\times10^8$ | | Wang et al. (2017) | Q | 80, 239 |
| JIJPRURLLPNXAY-UHFFFAOYSA-N | $2.5\times10^3$ | | Wang et al. (2017) | Q | 80, 240 |
| MCM:INB1HPCO3H | $1.5\times10^{12}$ | | Wang et al. (2017) | Q | 80, 238 |
| $C_5H_9NO_9$ | $1.3\times10^9$ | | Wang et al. (2017) | Q | 80, 239 |
| UQVCIXBCKWUTPD-UHFFFAOYSA-N | $2.2\times10^4$ | | Wang et al. (2017) | Q | 80, 240 |
| MCM:INB1HPPAN | $6.9\times10^{10}$ | | Wang et al. (2017) | Q | 80, 238 |
| $C_5H_8N_2O_{11}$ | $5.3\times10^8$ | | Wang et al. (2017) | Q | 80, 239 |
| MXDOILKWVFGWCS-UHFFFAOYSA-N | $1.6\times10^1$ | | Wang et al. (2017) | Q | 80, 240 |
| MCM:INB1NACO3H | $2.6\times10^9$ | | Wang et al. (2017) | Q | 80, 238 |
| $C_5H_8N_2O_{10}$ | $2.5\times10^8$ | | Wang et al. (2017) | Q | 80, 239 |
| IWCHPIGOQCZQCT-UHFFFAOYSA-N | $8.3$ | | Wang et al. (2017) | Q | 80, 240 |
| MCM:INB1NAPAN | $1.0\times10^8$ | | Wang et al. (2017) | Q | 80, 238 |
| $C_5H_7N_3O_{12}$ | $7.1\times10^7$ | | Wang et al. (2017) | Q | 80, 239 |
| NYNQOGZUUKXJDZ-UHFFFAOYSA-N | $2.8\times10^{-2}$ | | Wang et al. (2017) | Q | 80, 240 |
| MCM:INB1NBCO3H | $2.6\times10^9$ | | Wang et al. (2017) | Q | 80, 238 |
| $C_5H_8N_2O_{10}$ | $2.0\times10^8$ | | Wang et al. (2017) | Q | 80, 239 |
| KPBUCMMBNBINOH-UHFFFAOYSA-N | $3.5\times10^1$ | | Wang et al. (2017) | Q | 80, 240 |
| MCM:INB1NBPAN | $1.0\times10^8$ | | Wang et al. (2017) | Q | 80, 238 |
| $C_5H_7N_3O_{12}$ | $5.8\times10^7$ | | Wang et al. (2017) | Q | 80, 239 |
| JLOMTJWSNAHKLK-UHFFFAOYSA-N | $4.5\times10^{-2}$ | | Wang et al. (2017) | Q | 80, 240 |
| MCM:INB1NO3 | $7.8\times10^6$ | | Wang et al. (2017) | Q | 80, 238 |
| $C_5H_{10}N_2O_8$ | $1.4\times10^7$ | | Wang et al. (2017) | Q | 80, 239 |
| JJLGQNDDHJZYHP-UHFFFAOYSA-N | $8.1\times10^2$ | | Wang et al. (2017) | Q | 80, 240 |
| MCM:INB1OH | $6.9\times10^7$ | | Wang et al. (2017) | Q | 80, 238 |
| $C_5H_{11}NO_6$ | $1.7\times10^8$ | | Wang et al. (2017) | Q | 80, 239 |
| CMMIHJKAMSCCNX-UHFFFAOYSA-N | $1.1\times10^4$ | | Wang et al. (2017) | Q | 80, 240 |
| MCM:INB1OOH | $5.3\times10^9$ | | Wang et al. (2017) | Q | 80, 238 |
| $C_5H_{11}NO_7$ | $1.1\times10^8$ | | Wang et al. (2017) | Q | 80, 239 |
| QLIFJAFKYUVUMD-UHFFFAOYSA-N | $1.9\times10^5$ | | Wang et al. (2017) | Q | 80, 240 |
| MCM:INB2OOH | $5.3\times10^9$ | | Wang et al. (2017) | Q | 80, 238 |
| $C_5H_{11}NO_7$ | $5.6\times10^8$ | | Wang et al. (2017) | Q | 80, 239 |
| KNHLEPZPJYMJKZ-UHFFFAOYSA-N | $4.7\times10^4$ | | Wang et al. (2017) | Q | 80, 240 |



Table A4.6: Nitrates ($RONO_2$) (...continued)

| Substance Formula (Trivial Name) [CAS Registry Number] InChIKey | $H_s^{cp}$ (at $T^\ominus$) $\left[\dfrac{\text{mol}}{\text{m}^3\,\text{Pa}}\right]$ | $\dfrac{\text{d}\ln H_s^{cp}}{\text{d}(1/T)}$ [K] | Reference | Type | Note |
|---|---|---|---|---|---|
| MCM:INCNCO3H $C_5H_8N_2O_{10}$ OBBLXUYPBVASEH-UHFFFAOYSA-N | $2.7\times10^9$ $3.0\times10^8$ 8.0 | | Wang et al. (2017) Wang et al. (2017) Wang et al. (2017) | Q Q Q | 80, 238 80, 239 80, 240 |
| MCM:INCNO3 $C_5H_{10}N_2O_8$ YANCNBBJBURWKL-UHFFFAOYSA-N | $7.8\times10^6$ $2.7\times10^7$ $2.6\times10^2$ | | Wang et al. (2017) Wang et al. (2017) Wang et al. (2017) | Q Q Q | 80, 238 80, 239 80, 240 |
| MCM:INCNPAN $C_5H_7N_3O_{12}$ MINPDRSMAYIHHB-UHFFFAOYSA-N | $1.2\times10^8$ $5.4\times10^7$ $2.6\times10^{-2}$ | | Wang et al. (2017) Wang et al. (2017) Wang et al. (2017) | Q Q Q | 80, 238 80, 239 80, 240 |
| MCM:INCOH $C_5H_{11}NO_6$ CLYWBEAWDTVRBE-UHFFFAOYSA-N | $4.0\times10^7$ $2.8\times10^8$ $4.4\times10^3$ | | Wang et al. (2017) Wang et al. (2017) Wang et al. (2017) | Q Q Q | 80, 238 80, 239 80, 240 |
| MCM:INCOOH $C_5H_{11}NO_7$ LZOWWVNBCGLMRZ-UHFFFAOYSA-N | $5.3\times10^9$ $6.9\times10^8$ $2.0\times10^5$ | | Wang et al. (2017) Wang et al. (2017) Wang et al. (2017) | Q Q Q | 80, 238 80, 239 80, 240 |
| MCM:INDHCO3H $C_5H_9NO_8$ QUYOEJZVAFIYRP-UHFFFAOYSA-N | $3.5\times10^9$ $5.3\times10^7$ $5.1\times10^3$ | | Wang et al. (2017) Wang et al. (2017) Wang et al. (2017) | Q Q Q | 80, 238 80, 239 80, 240 |
| MCM:INDHPAN $C_5H_8N_2O_{10}$ WUJYJPIJVADQSX-UHFFFAOYSA-N | $1.4\times10^8$ $6.3\times10^8$ $3.7\times10^1$ | | Wang et al. (2017) Wang et al. (2017) Wang et al. (2017) | Q Q Q | 80, 238 80, 239 80, 240 |
| MCM:INDHPCO3H $C_5H_9NO_9$ XHWXDLVQLOECHL-UHFFFAOYSA-N | $1.5\times10^{12}$ $8.9\times10^8$ $6.5\times10^4$ | | Wang et al. (2017) Wang et al. (2017) Wang et al. (2017) | Q Q Q | 80, 238 80, 239 80, 240 |
| MCM:INDHPPAN $C_5H_8N_2O_{11}$ MNXVJBKQIAYCJC-UHFFFAOYSA-N | $6.9\times10^{10}$ $4.4\times10^8$ 7.8 | | Wang et al. (2017) Wang et al. (2017) Wang et al. (2017) | Q Q Q | 80, 238 80, 239 80, 240 |
| MCM:INDOH $C_5H_{11}NO_6$ HMAKIHHEIIANEM-UHFFFAOYSA-N | $6.9\times10^7$ $1.7\times10^8$ $3.7\times10^4$ | | Wang et al. (2017) Wang et al. (2017) Wang et al. (2017) | Q Q Q | 80, 238 80, 239 80, 240 |
| MCM:INDOOH $C_5H_{11}NO_7$ MAYHMSJZSHQSOH-UHFFFAOYSA-N | $5.3\times10^9$ $1.1\times10^8$ $7.1\times10^4$ | | Wang et al. (2017) Wang et al. (2017) Wang et al. (2017) | Q Q Q | 80, 238 80, 239 80, 240 |
| MCM:ISOPANO3 $C_5H_9NO_4$ ISDFXKLTMWDHJL-UHFFFAOYSA-N | $6.0\times10^1$ $1.9\times10^3$ $1.7\times10^1$ | | Wang et al. (2017) Wang et al. (2017) Wang et al. (2017) | Q Q Q | 80, 238 80, 239 80, 240 |
| MCM:ISOPBNO3 $C_5H_9NO_4$ CIXVZPWFWFMUOH-UHFFFAOYSA-N | $5.6\times10^1$ $1.1\times10^2$ $3.6\times10^{-1}$ | | Wang et al. (2017) Wang et al. (2017) Wang et al. (2017) | Q Q Q | 80, 238 80, 239 80, 240 |



Table A4.6: Nitrates ($RONO_2$) (. . . continued)

| Substance<br>Formula<br>(Trivial Name)<br>[CAS Registry Number]<br>InChIKey | $H_s^{cp}$<br>(at $T^\ominus$)<br>$\left[\dfrac{\mathrm{mol}}{\mathrm{m^3\,Pa}}\right]$ | $\dfrac{\mathrm{d}\ln H_s^{cp}}{\mathrm{d}(1/T)}$<br><br>[K] | Reference | Type | Note |
|---|---|---|---|---|---|
| MCM:ISOPCNO3 | $6.0\times10^1$ | | Wang et al. (2017) | Q | 80, 238 |
| $C_5H_9NO_4$ | $1.9\times10^3$ | | Wang et al. (2017) | Q | 80, 239 |
| IDJHVSOIJITEQU-UHFFFAOYSA-N | $2.3\times10^1$ | | Wang et al. (2017) | Q | 80, 240 |
| MCM:ISOPDNO3 | $5.9\times10^1$ | | Wang et al. (2017) | Q | 80, 238 |
| $C_5H_9NO_4$ | $1.8\times10^2$ | | Wang et al. (2017) | Q | 80, 239 |
| PYTOMGVARIWPAT-UHFFFAOYSA-N | 1.1 | | Wang et al. (2017) | Q | 80, 240 |
| MCM:M2BU2OLNO3 | $2.0\times10^1$ | | Wang et al. (2017) | Q | 80, 238 |
| $C_5H_{11}NO_4$ | $9.1\times10^1$ | | Wang et al. (2017) | Q | 80, 239 |
| AGMVBVSNQJHQAX-UHFFFAOYSA-N | $5.3\times10^{-1}$ | | Wang et al. (2017) | Q | 80, 240 |
| MCM:M2BUOL2NO3 | $1.7\times10^1$ | | Wang et al. (2017) | Q | 80, 238 |
| $C_5H_{11}NO_4$ | $7.4\times10^1$ | | Wang et al. (2017) | Q | 80, 239 |
| RQINUHXDGZGUIX-UHFFFAOYSA-N | $5.9\times10^{-1}$ | | Wang et al. (2017) | Q | 80, 240 |
| MCM:M3BU2OLNO3 | $3.4\times10^1$ | | Wang et al. (2017) | Q | 80, 238 |
| $C_5H_{11}NO_4$ | $1.4\times10^2$ | | Wang et al. (2017) | Q | 80, 239 |
| HJMHUJWXCZNPJL-UHFFFAOYSA-N | $3.2\times10^{-1}$ | | Wang et al. (2017) | Q | 80, 240 |
| MCM:MBOANO3 | $6.8\times10^4$ | | Wang et al. (2017) | Q | 80, 238 |
| $C_5H_{11}NO_5$ | $1.6\times10^5$ | | Wang et al. (2017) | Q | 80, 239 |
| SMFXIBRHTKUPLI-UHFFFAOYSA-N | $5.0\times10^2$ | | Wang et al. (2017) | Q | 80, 240 |
| MCM:MBOBNO3 | $2.1\times10^4$ | | Wang et al. (2017) | Q | 80, 238 |
| $C_5H_{11}NO_5$ | $1.6\times10^5$ | | Wang et al. (2017) | Q | 80, 239 |
| DYXSFFPMZZUQFC-UHFFFAOYSA-N | $4.5\times10^1$ | | Wang et al. (2017) | Q | 80, 240 |
| MCM:ME2BUOLNO3 | $2.0\times10^1$ | | Wang et al. (2017) | Q | 80, 238 |
| $C_5H_{11}NO_4$ | $8.1\times10^1$ | | Wang et al. (2017) | Q | 80, 239 |
| SBGDAJKUHMJEMT-UHFFFAOYSA-N | $4.8\times10^{-1}$ | | Wang et al. (2017) | Q | 80, 240 |
| MCM:ME3BUOLNO3 | $3.4\times10^1$ | | Wang et al. (2017) | Q | 80, 238 |
| $C_5H_{11}NO_4$ | $1.4\times10^2$ | | Wang et al. (2017) | Q | 80, 239 |
| YWTWLHUMBHVVJS-UHFFFAOYSA-N | 1.3 | | Wang et al. (2017) | Q | 80, 240 |
| MCM:NC4OHCO3H | $2.2\times10^7$ | | Wang et al. (2017) | Q | 80, 238 |
| $C_5H_9NO_7$ | $1.6\times10^6$ | | Wang et al. (2017) | Q | 80, 239 |
| PAYOKBFCVVFNAB-UHFFFAOYSA-N | 4.1 | | Wang et al. (2017) | Q | 80, 240 |
| MCM:NC4OHCPAN | $9.1\times10^5$ | | Wang et al. (2017) | Q | 80, 238 |
| $C_5H_8N_2O_9$ | $4.2\times10^5$ | | Wang et al. (2017) | Q | 80, 239 |
| JNBJPCXCCYUJJV-UHFFFAOYSA-N | $2.8\times10^{-2}$ | | Wang et al. (2017) | Q | 80, 240 |
| MCM:NC524NO3 | $2.7\times10^{10}$ | | Wang et al. (2017) | Q | 80, 238 |
| $C_5H_{10}N_2O_9$ | $1.6\times10^{10}$ | | Wang et al. (2017) | Q | 80, 239 |
| NSZZUBPWOYNZJW-UHFFFAOYSA-N | $1.0\times10^6$ | | Wang et al. (2017) | Q | 80, 240 |
| MCM:NC524OH | $1.1\times10^{11}$ | | Wang et al. (2017) | Q | 80, 238 |
| $C_5H_{11}NO_7$ | $1.4\times10^{11}$ | | Wang et al. (2017) | Q | 80, 239 |
| ZZGZKDXCMVGDSK-UHFFFAOYSA-N | $5.4\times10^6$ | | Wang et al. (2017) | Q | 80, 240 |





Table A4.6: Nitrates ($RONO_2$) (...continued)

| Substance Formula (Trivial Name) [CAS Registry Number] InChIKey | $H_s^{cp}$ (at $T^\ominus$) $\left[\dfrac{\text{mol}}{\text{m}^3\,\text{Pa}}\right]$ | $\dfrac{\text{d}\ln H_s^{cp}}{\text{d}(1/T)}$ [K] | Reference | Type | Note |
|---|---|---|---|---|---|
| MCM:NC524OOH $C_5H_{11}NO_8$ IVLBZHUBMDVGDP-UHFFFAOYSA-N | $1.8\times10^{13}$ $1.7\times10^{11}$ $9.3\times10^{5}$ | | Wang et al. (2017) Wang et al. (2017) Wang et al. (2017) | Q Q Q | 80, 238 80, 239 80, 240 |
| MCM:NMBOAOOH $C_5H_{11}NO_6$ JWWMAKNKCQEFPZ-UHFFFAOYSA-N | $1.5\times10^{6}$ $9.8\times10^{4}$ $1.0\times10^{3}$ | | Wang et al. (2017) Wang et al. (2017) Wang et al. (2017) | Q Q Q | 80, 238 80, 239 80, 240 |
| MCM:NMBOBOOH $C_5H_{11}NO_6$ ZVOCBPRLBFFNTM-UHFFFAOYSA-N | $1.5\times10^{6}$ $4.3\times10^{4}$ $3.8\times10^{2}$ | | Wang et al. (2017) Wang et al. (2017) Wang et al. (2017) | Q Q Q | 80, 238 80, 239 80, 240 |
| MCM:PE1ENEANO3 $C_5H_{11}NO_4$ VVLSYMLJBKJHQI-UHFFFAOYSA-N | $2.8\times10^{1}$ $1.1\times10^{2}$ $1.7$ | | Wang et al. (2017) Wang et al. (2017) Wang et al. (2017) | Q Q Q | 80, 238 80, 239 80, 240 |
| MCM:PE1ENEBNO3 $C_5H_{11}NO_4$ ZAYLQHAUCSLXCO-UHFFFAOYSA-N | $2.8\times10^{1}$ $1.3\times10^{2}$ $6.5\times10^{-1}$ | | Wang et al. (2017) Wang et al. (2017) Wang et al. (2017) | Q Q Q | 80, 238 80, 239 80, 240 |
| MCM:PE2ENEANO3 $C_5H_{11}NO_4$ SNSYWPGYFWSVQC-UHFFFAOYSA-N | $3.4\times10^{1}$ $1.2\times10^{2}$ $6.3\times10^{-1}$ | | Wang et al. (2017) Wang et al. (2017) Wang et al. (2017) | Q Q Q | 80, 238 80, 239 80, 240 |
| MCM:PE2ENEBNO3 $C_5H_{11}NO_4$ FSHUAFNCTFMDNJ-UHFFFAOYSA-N | $3.4\times10^{1}$ $1.1\times10^{2}$ $2.7\times10^{-1}$ | | Wang et al. (2017) Wang et al. (2017) Wang et al. (2017) | Q Q Q | 80, 238 80, 239 80, 240 |
| MCM:PROL11MNO3 $C_5H_{11}NO_4$ XLOGQOLFBMMDEL-UHFFFAOYSA-N | $1.7\times10^{1}$ $8.5\times10^{1}$ $5.4\times10^{-1}$ | | Wang et al. (2017) Wang et al. (2017) Wang et al. (2017) | Q Q Q | 80, 238 80, 239 80, 240 |
| MCM:PROL1MPAN $C_5H_9NO_6$ KGIFHBAEXLJKCU-UHFFFAOYSA-N | $3.0\times10^{2}$ $2.5\times10^{3}$ $1.5\times10^{-2}$ | | Wang et al. (2017) Wang et al. (2017) Wang et al. (2017) | Q Q Q | 80, 238 80, 239 80, 240 |
| MCM:BZBIPERNO3 $C_6H_7NO_6$ RBJPNJQUNYROJW-UHFFFAOYSA-N | $1.0\times10^{6}$ $8.7\times10^{4}$ $1.1\times10^{3}$ | | Wang et al. (2017) Wang et al. (2017) Wang et al. (2017) | Q Q Q | 80, 238 80, 239 80, 240 |
| MCM:C4ME2OHNO3 $C_6H_{13}NO_4$ KWGTWPROGIVFDL-UHFFFAOYSA-N | $1.1\times10^{1}$ $5.1\times10^{1}$ $1.7\times10^{-1}$ | | Wang et al. (2017) Wang et al. (2017) Wang et al. (2017) | Q Q Q | 80, 238 80, 239 80, 240 |
| MCM:C518PAN $C_6H_9NO_6$ SMGAVNKPBYTONP-UHFFFAOYSA-N | $1.5\times10^{4}$ $1.4\times10^{4}$ $4.1\times10^{-1}$ | | Wang et al. (2017) Wang et al. (2017) Wang et al. (2017) | Q Q Q | 80, 238 80, 239 80, 240 |
| MCM:C622NO3 $C_6H_{11}NO_4$ UCAFMJJTBHTENA-UHFFFAOYSA-N | $3.5\times10^{1}$ $4.0\times10^{1}$ $6.6\times10^{2}$ $2.0$ | 11000 | Wieser et al. (2023) Wang et al. (2017) Wang et al. (2017) Wang et al. (2017) | Q Q Q Q | 437 80, 238 80, 239 80, 240 |





Table A4.6: Nitrates ($RONO_2$) (... continued)

| Substance Formula (Trivial Name) [CAS Registry Number] InChIKey | $H_s^{cp}$ (at $T^\ominus$) $\left[\dfrac{\mathrm{mol}}{\mathrm{m}^3\,\mathrm{Pa}}\right]$ | $\dfrac{\mathrm{d}\ln H_s^{cp}}{\mathrm{d}(1/T)}$ [K] | Reference | Type | Note |
|---|---|---|---|---|---|
| MCM:C624NO3 | $4.0\times10^1$ | | Wang et al. (2017) | Q | 80, 238 |
| $C_6H_{11}NO_4$ | $3.8\times10^2$ | | Wang et al. (2017) | Q | 80, 239 |
| NPRBCQJDIIBUMY-UHFFFAOYSA-N | 5.4 | | Wang et al. (2017) | Q | 80, 240 |
| MCM:C64OH5NO3 | $2.7\times10^1$ | | Wang et al. (2017) | Q | 80, 238 |
| $C_6H_{13}NO_4$ | $7.8\times10^1$ | | Wang et al. (2017) | Q | 80, 239 |
| SZVWRLIYDKPIRS-UHFFFAOYSA-N | $2.6\times10^{-1}$ | | Wang et al. (2017) | Q | 80, 240 |
| MCM:C65OH4NO3 | $2.7\times10^1$ | | Wang et al. (2017) | Q | 80, 238 |
| $C_6H_{13}NO_4$ | $8.7\times10^1$ | | Wang et al. (2017) | Q | 80, 239 |
| ZZPOFDLVCBSRPC-UHFFFAOYSA-N | $5.5\times10^{-1}$ | | Wang et al. (2017) | Q | 80, 240 |
| MCM:C66NO35OH | $2.3\times10^1$ | | Wang et al. (2017) | Q | 80, 238 |
| $C_6H_{13}NO_4$ | $8.9\times10^1$ | | Wang et al. (2017) | Q | 80, 239 |
| FGVUHVRIXFWTDQ-UHFFFAOYSA-N | $5.9\times10^{-1}$ | | Wang et al. (2017) | Q | 80, 240 |
| MCM:C6OH5NO3 | $2.3\times10^1$ | | Wang et al. (2017) | Q | 80, 238 |
| $C_6H_{13}NO_4$ | $8.1\times10^1$ | | Wang et al. (2017) | Q | 80, 239 |
| WNZDORMXFCCFFY-UHFFFAOYSA-N | $8.3\times10^{-1}$ | | Wang et al. (2017) | Q | 80, 240 |
| MCM:C6PAN11 | $7.8\times10^3$ | | Wang et al. (2017) | Q | 80, 238 |
| $C_6H_{11}NO_6$ | $2.8\times10^4$ | | Wang et al. (2017) | Q | 80, 239 |
| MMZOJEONIKVQSZ-UHFFFAOYSA-N | $6.9\times10^{-1}$ | | Wang et al. (2017) | Q | 80, 240 |
| MCM:C6PAN13 | $4.5\times10^3$ | | Wang et al. (2017) | Q | 80, 238 |
| $C_6H_{11}NO_6$ | $1.8\times10^4$ | | Wang et al. (2017) | Q | 80, 239 |
| UVFJOYDNGFNBNV-UHFFFAOYSA-N | 1.0 | | Wang et al. (2017) | Q | 80, 240 |
| MCM:C6PAN14 | $4.5\times10^3$ | | Wang et al. (2017) | Q | 80, 238 |
| $C_6H_{11}NO_6$ | $1.9\times10^4$ | | Wang et al. (2017) | Q | 80, 239 |
| OFISHWPBQMHPEB-UHFFFAOYSA-N | $1.4\times10^{-1}$ | | Wang et al. (2017) | Q | 80, 240 |
| MCM:C6PAN19 | $7.8\times10^3$ | | Wang et al. (2017) | Q | 80, 238 |
| $C_6H_{11}NO_6$ | $2.6\times10^4$ | | Wang et al. (2017) | Q | 80, 239 |
| ZTVAXWHNQKLGPW-UHFFFAOYSA-N | 1.2 | | Wang et al. (2017) | Q | 80, 240 |
| MCM:C6PAN1 | $6.5\times10^3$ | | Wang et al. (2017) | Q | 80, 238 |
| $C_6H_{11}NO_6$ | $2.6\times10^4$ | | Wang et al. (2017) | Q | 80, 239 |
| INIWXNWKFGUMOJ-UHFFFAOYSA-N | $6.2\times10^{-1}$ | | Wang et al. (2017) | Q | 80, 240 |
| MCM:C6PAN20 | $7.8\times10^3$ | | Wang et al. (2017) | Q | 80, 238 |
| $C_6H_{11}NO_6$ | $1.5\times10^4$ | | Wang et al. (2017) | Q | 80, 239 |
| HBAXDJNNSYEITK-UHFFFAOYSA-N | $4.8\times10^{-2}$ | | Wang et al. (2017) | Q | 80, 240 |
| MCM:C6PAN21 | $4.5\times10^3$ | | Wang et al. (2017) | Q | 80, 238 |
| $C_6H_{11}NO_6$ | $1.6\times10^4$ | | Wang et al. (2017) | Q | 80, 239 |
| PXGXPDBWNDXFTJ-UHFFFAOYSA-N | $3.0\times10^{-2}$ | | Wang et al. (2017) | Q | 80, 240 |
| MCM:C6PAN22 | $6.5\times10^3$ | | Wang et al. (2017) | Q | 80, 238 |
| $C_6H_{11}NO_6$ | $1.3\times10^4$ | | Wang et al. (2017) | Q | 80, 239 |
| YIFZDOMLXOKIHL-UHFFFAOYSA-N | $1.1\times10^{-1}$ | | Wang et al. (2017) | Q | 80, 240 |





Table A4.6: Nitrates ($RONO_2$) (...continued)

| Substance Formula (Trivial Name) [CAS Registry Number] InChIKey | $H_s^{cp}$ (at $T^{\ominus}$) $\left[\dfrac{\text{mol}}{\text{m}^3\,\text{Pa}}\right]$ | $\dfrac{\text{d}\ln H_s^{cp}}{\text{d}(1/T)}$ [K] | Reference | Type | Note |
|---|---|---|---|---|---|
| MCM:C6PAN23 | $3.3\times10^2$ | | Wang et al. (2017) | Q | 80, 238 |
| $C_6H_{11}NO_6$ | $2.6\times10^3$ | | Wang et al. (2017) | Q | 80, 239 |
| OODNPGDDVDMKSB-UHFFFAOYSA-N | $4.2\times10^{-2}$ | | Wang et al. (2017) | Q | 80, 240 |
| MCM:C6PAN4 | $7.8\times10^3$ | | Wang et al. (2017) | Q | 80, 238 |
| $C_6H_{11}NO_6$ | $2.1\times10^4$ | | Wang et al. (2017) | Q | 80, 239 |
| ZWWUNLHZACGCEV-UHFFFAOYSA-N | 1.1 | | Wang et al. (2017) | Q | 80, 240 |
| MCM:C6PAN8 | $4.5\times10^3$ | | Wang et al. (2017) | Q | 80, 238 |
| $C_6H_{11}NO_6$ | $2.8\times10^4$ | | Wang et al. (2017) | Q | 80, 239 |
| UOLRWSMDFSCHJH-UHFFFAOYSA-N | $7.1\times10^{-1}$ | | Wang et al. (2017) | Q | 80, 240 |
| MCM:CYHXOLANO3 | $8.5\times10^1$ | | Wang et al. (2017) | Q | 80, 238 |
| $C_6H_{11}NO_4$ | $4.6\times10^2$ | | Wang et al. (2017) | Q | 80, 239 |
| GVSCAKVHXWAQQJ-UHFFFAOYSA-N | 4.7 | | Wang et al. (2017) | Q | 80, 240 |
| MCM:H13M3C5NO3 | $4.6\times10^4$ | | Wang et al. (2017) | Q | 80, 238 |
| $C_6H_{13}NO_5$ | $2.8\times10^5$ | | Wang et al. (2017) | Q | 80, 239 |
| ZRQJKBICSYYJDC-UHFFFAOYSA-N | $4.2\times10^2$ | | Wang et al. (2017) | Q | 80, 240 |
| MCM:H1MC5NO3 | $1.9\times10^1$ | | Wang et al. (2017) | Q | 80, 238 |
| $C_6H_{13}NO_4$ | $3.6\times10^2$ | | Wang et al. (2017) | Q | 80, 239 |
| YQESDLQNDPCVPT-UHFFFAOYSA-N | 8.7 | | Wang et al. (2017) | Q | 80, 240 |
| MCM:H2MC5NO3 | $1.9\times10^1$ | | Wang et al. (2017) | Q | 80, 238 |
| $C_6H_{13}NO_4$ | $4.3\times10^2$ | | Wang et al. (2017) | Q | 80, 239 |
| OQOPDPWKHFKGPO-UHFFFAOYSA-N | 4.7 | | Wang et al. (2017) | Q | 80, 240 |
| MCM:H3M3C5NO3 | $1.2\times10^1$ | | Wang et al. (2017) | Q | 80, 238 |
| $C_6H_{13}NO_4$ | $2.5\times10^2$ | | Wang et al. (2017) | Q | 80, 239 |
| KFPWVQORUDKTSF-UHFFFAOYSA-N | 1.1 | | Wang et al. (2017) | Q | 80, 240 |
| MCM:HM22C4NO3 | $1.2\times10^1$ | | Wang et al. (2017) | Q | 80, 238 |
| $C_6H_{13}NO_4$ | $3.0\times10^2$ | | Wang et al. (2017) | Q | 80, 239 |
| VALJBIPPHZOXME-UHFFFAOYSA-N | 1.1 | | Wang et al. (2017) | Q | 80, 240 |
| MCM:HM23C4NO3 | $1.9\times10^1$ | | Wang et al. (2017) | Q | 80, 238 |
| $C_6H_{13}NO_4$ | $4.8\times10^2$ | | Wang et al. (2017) | Q | 80, 239 |
| VYFSYIPQAVXJDI-UHFFFAOYSA-N | 4.5 | | Wang et al. (2017) | Q | 80, 240 |
| MCM:HM33C4NO3 | $1.2\times10^1$ | | Wang et al. (2017) | Q | 80, 238 |
| $C_6H_{13}NO_4$ | $3.2\times10^2$ | | Wang et al. (2017) | Q | 80, 239 |
| HUEIARDLQBFSAX-UHFFFAOYSA-N | 4.9 | | Wang et al. (2017) | Q | 80, 240 |
| MCM:HO1C6NO3 | $1.9\times10^1$ | | Wang et al. (2017) | Q | 80, 238 |
| $C_6H_{13}NO_4$ | $2.5\times10^2$ | | Wang et al. (2017) | Q | 80, 239 |
| ITCMBKDCMKPMFX-UHFFFAOYSA-N | 7.8 | | Wang et al. (2017) | Q | 80, 240 |
| MCM:HO1MC5NO3 | $1.9\times10^1$ | | Wang et al. (2017) | Q | 80, 238 |
| $C_6H_{13}NO_4$ | $2.4\times10^2$ | | Wang et al. (2017) | Q | 80, 239 |
| HKEZUADWXQLOIL-UHFFFAOYSA-N | 4.9 | | Wang et al. (2017) | Q | 80, 240 |





Table A4.6: Nitrates ($RONO_2$) (...continued)

| Substance<br>Formula<br>(Trivial Name)<br>[CAS Registry Number]<br>InChIKey | $H_s^{cp}$<br>(at $T^{\ominus}$)<br>$\left[\dfrac{\text{mol}}{\text{m}^3\,\text{Pa}}\right]$ | $\dfrac{\text{d}\ln H_s^{cp}}{\text{d}(1/T)}$<br><br>[K] | Reference | Type | Note |
|---|---|---|---|---|---|
| MCM:HO2C6NO3 | $1.6\times10^1$ | 10000 | Wieser et al. (2023) | Q | 437 |
| $C_6H_{13}NO_4$ | $1.9\times10^1$ | | Wang et al. (2017) | Q | 80, 238 |
| PAWWQEMXBSLVJC-UHFFFAOYSA-N | $2.0\times10^2$ | | Wang et al. (2017) | Q | 80, 239 |
| | 4.2 | | Wang et al. (2017) | Q | 80, 240 |
| MCM:HO2M2C5NO3 | $1.2\times10^1$ | | Wang et al. (2017) | Q | 80, 238 |
| $C_6H_{13}NO_4$ | $2.9\times10^2$ | | Wang et al. (2017) | Q | 80, 239 |
| VYPLOHRVLBPDEY-UHFFFAOYSA-N | 5.0 | | Wang et al. (2017) | Q | 80, 240 |
| MCM:HO2MC5NO3 | $1.9\times10^1$ | | Wang et al. (2017) | Q | 80, 238 |
| $C_6H_{13}NO_4$ | $3.1\times10^2$ | | Wang et al. (2017) | Q | 80, 239 |
| HYCOTEZKEHIINQ-UHFFFAOYSA-N | 3.5 | | Wang et al. (2017) | Q | 80, 240 |
| MCM:HO3C6NO3 | $1.9\times10^1$ | | Wang et al. (2017) | Q | 80, 238 |
| $C_6H_{13}NO_4$ | $3.0\times10^2$ | | Wang et al. (2017) | Q | 80, 239 |
| DLCRGHMYOOYQLW-UHFFFAOYSA-N | 2.8 | | Wang et al. (2017) | Q | 80, 240 |
| MCM:NDNPHENOOH | $1.2\times10^{15}$ | | Wang et al. (2017) | Q | 80, 238 |
| $C_6H_5N_3O_{12}$ | $3.2\times10^9$ | | Wang et al. (2017) | Q | 80, 239 |
| CXGWHQULUCBXTG-UHFFFAOYSA-N | $1.8\times10^3$ | | Wang et al. (2017) | Q | 80, 240 |
| MCM:NNCATECOOH | $1.4\times10^{16}$ | | Wang et al. (2017) | Q | 80, 238 |
| $C_6H_6N_2O_{11}$ | $5.9\times10^8$ | | Wang et al. (2017) | Q | 80, 239 |
| SEZKEDQZGXKDIE-UHFFFAOYSA-N | $1.0\times10^7$ | | Wang et al. (2017) | Q | 80, 240 |
| MCM:NPHENOH | $1.8\times10^9$ | | Wang et al. (2017) | Q | 80, 238 |
| $C_6H_7NO_7$ | $8.5\times10^6$ | | Wang et al. (2017) | Q | 80, 239 |
| RMHIEAGXGVAIOV-UHFFFAOYSA-N | $4.8\times10^5$ | | Wang et al. (2017) | Q | 80, 240 |
| MCM:NPHENOOH | $4.7\times10^{10}$ | | Wang et al. (2017) | Q | 80, 238 |
| $C_6H_7NO_8$ | $5.8\times10^6$ | | Wang et al. (2017) | Q | 80, 239 |
| AMKSGTJAMDYUMG-UHFFFAOYSA-N | $4.3\times10^4$ | | Wang et al. (2017) | Q | 80, 240 |
| MCM:C622PAN | $1.3\times10^4$ | | Wang et al. (2017) | Q | 80, 238 |
| $C_7H_{11}NO_6$ | $2.8\times10^4$ | | Wang et al. (2017) | Q | 80, 239 |
| YZCFVDKMDQZEJM-UHFFFAOYSA-N | 1.1 | | Wang et al. (2017) | Q | 80, 240 |
| MCM:C624PAN | $1.3\times10^4$ | | Wang et al. (2017) | Q | 80, 238 |
| $C_7H_{11}NO_6$ | $2.0\times10^4$ | | Wang et al. (2017) | Q | 80, 239 |
| GVGFFJILZZBAQC-UHFFFAOYSA-N | 2.3 | | Wang et al. (2017) | Q | 80, 240 |
| MCM:C720NO3 | $1.6\times10^2$ | | Wang et al. (2017) | Q | 80, 238 |
| $C_7H_{11}NO_4$ | $1.7\times10^3$ | | Wang et al. (2017) | Q | 80, 239 |
| VRUSDNGIDALYSE-UHFFFAOYSA-N | $3.0\times10^1$ | | Wang et al. (2017) | Q | 80, 240 |
| MCM:C7PAN1 | $3.7\times10^3$ | | Wang et al. (2017) | Q | 80, 238 |
| $C_7H_{13}NO_6$ | $2.1\times10^4$ | | Wang et al. (2017) | Q | 80, 239 |
| HWQKKNUBUUPBAM-UHFFFAOYSA-N | $2.8\times10^{-1}$ | | Wang et al. (2017) | Q | 80, 240 |
| MCM:C7PAN2 | $6.0\times10^3$ | | Wang et al. (2017) | Q | 80, 238 |
| $C_7H_{13}NO_6$ | $9.8\times10^3$ | | Wang et al. (2017) | Q | 80, 239 |
| LEPHBBVBSNQAKS-UHFFFAOYSA-N | $6.9\times10^{-2}$ | | Wang et al. (2017) | Q | 80, 240 |





Table A4.6: Nitrates ($RONO_2$) (...continued)

| Substance Formula (Trivial Name) [CAS Registry Number] InChIKey | $H_s^{cp}$ (at $T^\ominus$) $\left[\dfrac{\mathrm{mol}}{\mathrm{m^3\,Pa}}\right]$ | $\dfrac{\mathrm{d}\ln H_s^{cp}}{\mathrm{d}(1/T)}$ [K] | Reference | Type | Note |
|---|---|---|---|---|---|
| MCM:H2M2C65NO3 | $1.0\times10^1$ | | Wang et al. (2017) | Q | 80, 238 |
| $C_7H_{15}NO_4$ | $1.4\times10^2$ | | Wang et al. (2017) | Q | 80, 239 |
| CAEJQFHLOCBFOE-UHFFFAOYSA-N | 3.7 | | Wang et al. (2017) | Q | 80, 240 |
| MCM:H2M4C65NO3 | $1.8\times10^1$ | | Wang et al. (2017) | Q | 80, 238 |
| $C_7H_{15}NO_4$ | $2.1\times10^2$ | | Wang et al. (2017) | Q | 80, 239 |
| GNQNFXQMHRYYMA-UHFFFAOYSA-N | 3.2 | | Wang et al. (2017) | Q | 80, 240 |
| MCM:H2M5C65NO3 | $1.0\times10^1$ | | Wang et al. (2017) | Q | 80, 238 |
| $C_7H_{15}NO_4$ | $9.6\times10^1$ | | Wang et al. (2017) | Q | 80, 239 |
| JWGQNCMAKPVWAL-UHFFFAOYSA-N | $1.6\times10^1$ | | Wang et al. (2017) | Q | 80, 240 |
| MCM:H3M3C6NO3 | $1.0\times10^1$ | | Wang et al. (2017) | Q | 80, 238 |
| $C_7H_{15}NO_4$ | $2.3\times10^2$ | | Wang et al. (2017) | Q | 80, 239 |
| HWCRTHIHVAVLJX-UHFFFAOYSA-N | 2.3 | | Wang et al. (2017) | Q | 80, 240 |
| MCM:HO3C76NO3 | $1.7\times10^1$ | | Wang et al. (2017) | Q | 80, 238 |
| $C_7H_{15}NO_4$ | $1.5\times10^2$ | | Wang et al. (2017) | Q | 80, 239 |
| FLDFLEPRHCTYNR-UHFFFAOYSA-N | 2.0 | | Wang et al. (2017) | Q | 80, 240 |
| MCM:MNNCATCOOH | $1.0\times10^{16}$ | | Wang et al. (2017) | Q | 80, 238 |
| $C_7H_8N_2O_{11}$ | $5.5\times10^9$ | | Wang et al. (2017) | Q | 80, 239 |
| QARZHCGSBCBWKI-UHFFFAOYSA-N | $1.2\times10^5$ | | Wang et al. (2017) | Q | 80, 240 |
| MCM:NCRESOH | $9.8\times10^8$ | | Wang et al. (2017) | Q | 80, 238 |
| $C_7H_9NO_7$ | $3.4\times10^6$ | | Wang et al. (2017) | Q | 80, 239 |
| DNQMNVUMZARJRH-UHFFFAOYSA-N | $7.3\times10^5$ | | Wang et al. (2017) | Q | 80, 240 |
| MCM:NCRESOOH | $2.6\times10^{10}$ | | Wang et al. (2017) | Q | 80, 238 |
| $C_7H_9NO_8$ | $2.1\times10^6$ | | Wang et al. (2017) | Q | 80, 239 |
| CBYLLFTZDVJCIH-UHFFFAOYSA-N | $6.9\times10^4$ | | Wang et al. (2017) | Q | 80, 240 |
| MCM:NDNCRESOOH | $7.1\times10^{14}$ | | Wang et al. (2017) | Q | 80, 238 |
| $C_7H_7N_3O_{12}$ | $1.4\times10^9$ | | Wang et al. (2017) | Q | 80, 239 |
| LRLYMWFOIDHJSH-UHFFFAOYSA-N | $1.3\times10^3$ | | Wang et al. (2017) | Q | 80, 240 |
| MCM:TLBIPERNO3 | $5.8\times10^5$ | | Wang et al. (2017) | Q | 80, 238 |
| $C_7H_9NO_6$ | $2.5\times10^4$ | | Wang et al. (2017) | Q | 80, 239 |
| BBVQCFWACDVWPL-UHFFFAOYSA-N | $4.9\times10^2$ | | Wang et al. (2017) | Q | 80, 240 |
| MCM:C8PAN1 | $4.7\times10^3$ | | Wang et al. (2017) | Q | 80, 238 |
| $C_8H_{15}NO_6$ | $8.0\times10^3$ | | Wang et al. (2017) | Q | 80, 239 |
| JRARROZAJMGAPP-UHFFFAOYSA-N | $7.3\times10^{-2}$ | | Wang et al. (2017) | Q | 80, 240 |
| MCM:EBZBPERNO3 | $5.4\times10^5$ | | Wang et al. (2017) | Q | 80, 238 |
| $C_8H_{11}NO_6$ | $1.6\times10^4$ | | Wang et al. (2017) | Q | 80, 239 |
| GVVNXZZYBMLIBS-UHFFFAOYSA-N | $1.9\times10^2$ | | Wang et al. (2017) | Q | 80, 240 |
| MCM:ENNCATCOOH | $8.7\times10^{15}$ | | Wang et al. (2017) | Q | 80, 238 |
| $C_8H_{10}N_2O_{11}$ | $3.2\times10^9$ | | Wang et al. (2017) | Q | 80, 239 |
| SUCBZAKUUVBJMN-UHFFFAOYSA-N | $5.8\times10^4$ | | Wang et al. (2017) | Q | 80, 240 |





Table A4.6: Nitrates ($RONO_2$) (... continued)

| Substance<br>Formula<br>(Trivial Name)<br>[CAS Registry Number]<br>InChIKey | $H_s^{cp}$<br>(at $T^\ominus$)<br>$\left[\dfrac{\text{mol}}{\text{m}^3\,\text{Pa}}\right]$ | $\dfrac{\text{d}\ln H_s^{cp}}{\text{d}(1/T)}$<br><br>[K] | Reference | Type | Note |
|---|---|---|---|---|---|
| MCM:HO3C86NO3<br>$C_8H_{17}NO_4$<br>WNECKFLKAYSMQF-UHFFFAOYSA-N | $1.4\times10^1$<br>$1.2\times10^2$<br>$1.8$ | | Wang et al. (2017)<br>Wang et al. (2017)<br>Wang et al. (2017) | Q<br>Q<br>Q | 80, 238<br>80, 239<br>80, 240 |
| MCM:MXNNCATOOH<br>$C_8H_{10}N_2O_{11}$<br>SNJAPGUPFXVMJK-UHFFFAOYSA-N | $5.4\times10^{15}$<br>$2.0\times10^9$<br>$4.9\times10^4$ | | Wang et al. (2017)<br>Wang et al. (2017)<br>Wang et al. (2017) | Q<br>Q<br>Q | 80, 238<br>80, 239<br>80, 240 |
| MCM:MXYBIPENO3<br>$C_8H_{11}NO_6$<br>QFCKPTYGEYQVET-UHFFFAOYSA-N | $3.3\times10^5$<br>$8.9\times10^3$<br>$1.3\times10^2$ | | Wang et al. (2017)<br>Wang et al. (2017)<br>Wang et al. (2017) | Q<br>Q<br>Q | 80, 238<br>80, 239<br>80, 240 |
| MCM:NDNEBZLOOH<br>$C_8H_9N_3O_{12}$<br>XZASAIPCUIAVQG-UHFFFAOYSA-N | $6.3\times10^{14}$<br>$8.5\times10^8$<br>$4.4\times10^2$ | | Wang et al. (2017)<br>Wang et al. (2017)<br>Wang et al. (2017) | Q<br>Q<br>Q | 80, 238<br>80, 239<br>80, 240 |
| MCM:NDNMXYLOOH<br>$C_8H_9N_3O_{12}$<br>AAXPRTISYYDKLT-UHFFFAOYSA-N | $4.6\times10^{14}$<br>$8.0\times10^9$<br>$1.9\times10^4$ | | Wang et al. (2017)<br>Wang et al. (2017)<br>Wang et al. (2017) | Q<br>Q<br>Q | 80, 238<br>80, 239<br>80, 240 |
| MCM:NDNOXYLOOH<br>$C_8H_9N_3O_{12}$<br>SEMBUWBKKSEFTL-UHFFFAOYSA-N | $8.5\times10^{14}$<br>$2.3\times10^9$<br>$9.1\times10^3$ | | Wang et al. (2017)<br>Wang et al. (2017)<br>Wang et al. (2017) | Q<br>Q<br>Q | 80, 238<br>80, 239<br>80, 240 |
| MCM:NDNPXYLOOH<br>$C_8H_9N_3O_{12}$<br>YXONGYHUJBRFJO-UHFFFAOYSA-N | $4.6\times10^{14}$<br>$1.2\times10^9$<br>$4.0\times10^2$ | | Wang et al. (2017)<br>Wang et al. (2017)<br>Wang et al. (2017) | Q<br>Q<br>Q | 80, 238<br>80, 239<br>80, 240 |
| MCM:NEBNZOLOH<br>$C_8H_{11}NO_7$<br>GKEIZOZEJLMBMT-UHFFFAOYSA-N | $7.8\times10^8$<br>$2.1\times10^6$<br>$2.3\times10^5$ | | Wang et al. (2017)<br>Wang et al. (2017)<br>Wang et al. (2017) | Q<br>Q<br>Q | 80, 238<br>80, 239<br>80, 240 |
| MCM:NEBNZOLOOH<br>$C_8H_{11}NO_8$<br>DZFBJORWJYJOSR-UHFFFAOYSA-N | $2.0\times10^{10}$<br>$1.3\times10^6$<br>$3.9\times10^4$ | | Wang et al. (2017)<br>Wang et al. (2017)<br>Wang et al. (2017) | Q<br>Q<br>Q | 80, 238<br>80, 239<br>80, 240 |
| MCM:NMXYOLOH<br>$C_8H_{11}NO_7$<br>SFEIDLQIDXHPNL-UHFFFAOYSA-N | $5.4\times10^8$<br>$1.4\times10^6$<br>$3.6\times10^4$ | | Wang et al. (2017)<br>Wang et al. (2017)<br>Wang et al. (2017) | Q<br>Q<br>Q | 80, 238<br>80, 239<br>80, 240 |
| MCM:NMXYOLOOH<br>$C_8H_{11}NO_8$<br>ZTRVHLWFIBLUPT-UHFFFAOYSA-N | $1.4\times10^{10}$<br>$8.1\times10^5$<br>$6.6\times10^4$ | | Wang et al. (2017)<br>Wang et al. (2017)<br>Wang et al. (2017) | Q<br>Q<br>Q | 80, 238<br>80, 239<br>80, 240 |
| MCM:NOXYOLOH<br>$C_8H_{11}NO_7$<br>UBRWKHDWLQNZCI-UHFFFAOYSA-N | $6.6\times10^8$<br>$2.3\times10^6$<br>$1.0\times10^5$ | | Wang et al. (2017)<br>Wang et al. (2017)<br>Wang et al. (2017) | Q<br>Q<br>Q | 80, 238<br>80, 239<br>80, 240 |
| MCM:NOXYOLOOH<br>$C_8H_{11}NO_8$<br>UHJNRWMXKFNWGQ-UHFFFAOYSA-N | $1.7\times10^{10}$<br>$1.5\times10^6$<br>$2.0\times10^4$ | | Wang et al. (2017)<br>Wang et al. (2017)<br>Wang et al. (2017) | Q<br>Q<br>Q | 80, 238<br>80, 239<br>80, 240 |





Table A4.6: Nitrates ($RONO_2$) (. . . continued)

| Substance / Formula / (Trivial Name) / [CAS Registry Number] / InChIKey | $H_s^{cp}$ (at $T^\ominus$) $\left[\dfrac{\text{mol}}{\text{m}^3\,\text{Pa}}\right]$ | $\dfrac{\mathrm{d}\ln H_s^{cp}}{\mathrm{d}(1/T)}$ [K] | Reference | Type | Note |
|---|---|---|---|---|---|
| MCM:NPXYOLOH $C_8H_{11}NO_7$ DMOUSJNYQFTOEV-UHFFFAOYSA-N | $6.6\times10^8$ $3.0\times10^6$ $2.6\times10^4$ | | Wang et al. (2017) Wang et al. (2017) Wang et al. (2017) | Q Q Q | 80, 238 80, 239 80, 240 |
| MCM:NPXYOLOOH $C_8H_{11}NO_8$ XNVGITFEYRRJBZ-UHFFFAOYSA-N | $1.7\times10^{10}$ $1.8\times10^6$ $3.0\times10^4$ | | Wang et al. (2017) Wang et al. (2017) Wang et al. (2017) | Q Q Q | 80, 238 80, 239 80, 240 |
| MCM:OXNNCATOOH $C_8H_{10}N_2O_{11}$ GYUDSTHOULZHIU-UHFFFAOYSA-N | $1.0\times10^{16}$ $3.5\times10^8$ $3.2\times10^4$ | | Wang et al. (2017) Wang et al. (2017) Wang et al. (2017) | Q Q Q | 80, 238 80, 239 80, 240 |
| MCM:OXYBIPENO3 $C_8H_{11}NO_6$ QQSYVIZUYAXYBZ-UHFFFAOYSA-N | $3.3\times10^5$ $9.1\times10^3$ $2.0\times10^4$ | | Wang et al. (2017) Wang et al. (2017) Wang et al. (2017) | Q Q Q | 80, 238 80, 239 80, 240 |
| MCM:PXNNCATOOH $C_8H_{10}N_2O_{11}$ PMLOPKQQVRFHBT-UHFFFAOYSA-N | $5.4\times10^{15}$ $2.0\times10^8$ $4.5\times10^4$ | | Wang et al. (2017) Wang et al. (2017) Wang et al. (2017) | Q Q Q | 80, 238 80, 239 80, 240 |
| MCM:PXYBIPENO3 $C_8H_{11}NO_6$ DHJHBQWQGWNRCX-UHFFFAOYSA-N | $3.3\times10^5$ $9.3\times10^3$ $2.9\times10^2$ | | Wang et al. (2017) Wang et al. (2017) Wang et al. (2017) | Q Q Q | 80, 238 80, 239 80, 240 |
| MCM:C9PAN1 $C_9H_{17}NO_6$ XXDOIRAKHRPRCF-UHFFFAOYSA-N | $3.7\times10^3$ $6.8\times10^3$ $6.3\times10^{-2}$ | | Wang et al. (2017) Wang et al. (2017) Wang et al. (2017) | Q Q Q | 80, 238 80, 239 80, 240 |
| MCM:HO3C96NO3 $C_9H_{19}NO_4$ IBTJZAOBCFQTAU-UHFFFAOYSA-N | $1.1\times10^1$ $9.3\times10^1$ $1.1$ | | Wang et al. (2017) Wang et al. (2017) Wang et al. (2017) | Q Q Q | 80, 238 80, 239 80, 240 |
| MCM:IPBZBPRNO3 $C_9H_{13}NO_6$ VLTHUNWGYHRCPS-UHFFFAOYSA-N | $5.0\times10^5$ $1.5\times10^4$ $1.2\times10^2$ | | Wang et al. (2017) Wang et al. (2017) Wang et al. (2017) | Q Q Q | 80, 238 80, 239 80, 240 |
| MCM:IPNNCATOOH $C_9H_{12}N_2O_{11}$ QCQAPHFHFZPQHQ-UHFFFAOYSA-N | $8.1\times10^{15}$ $2.9\times10^9$ $1.4\times10^4$ | | Wang et al. (2017) Wang et al. (2017) Wang et al. (2017) | Q Q Q | 80, 238 80, 239 80, 240 |
| MCM:METLBIPNO3 $C_9H_{13}NO_6$ VQZACYIDHNLNBQ-UHFFFAOYSA-N | $2.6\times10^5$ $5.9\times10^3$ $2.8\times10^1$ | | Wang et al. (2017) Wang et al. (2017) Wang et al. (2017) | Q Q Q | 80, 238 80, 239 80, 240 |
| MCM:MTNNCATOOH $C_9H_{12}N_2O_{11}$ ILPSXJBNDMCAEH-UHFFFAOYSA-N | $4.8\times10^{15}$ $1.2\times10^9$ $9.3\times10^3$ | | Wang et al. (2017) Wang et al. (2017) Wang et al. (2017) | Q Q Q | 80, 238 80, 239 80, 240 |
| MCM:NDNIPBLOOH $C_9H_{11}N_3O_{12}$ HEZPBZRCUHPAHJ-UHFFFAOYSA-N | $5.9\times10^{14}$ $8.0\times10^8$ $2.8\times10^2$ | | Wang et al. (2017) Wang et al. (2017) Wang et al. (2017) | Q Q Q | 80, 238 80, 239 80, 240 |





Table A4.6: Nitrates ($RONO_2$) (...continued)

| Substance Formula (Trivial Name) [CAS Registry Number] InChIKey | $H_s^{cp}$ (at $T^\ominus$) $\left[\dfrac{\mathrm{mol}}{\mathrm{m^3\,Pa}}\right]$ | $\dfrac{\mathrm{d}\ln H_s^{cp}}{\mathrm{d}(1/T)}$ [K] | Reference | Type | Note |
|---|---|---|---|---|---|
| MCM:NDNMETLOOH | $3.8\times10^{14}$ | | Wang et al. (2017) | Q | 80, 238 |
| $C_9H_{11}N_3O_{12}$ | $4.9\times10^{9}$ | | Wang et al. (2017) | Q | 80, 239 |
| GVOHWVMFRUNQQB-UHFFFAOYSA-N | $1.1\times10^{3}$ | | Wang et al. (2017) | Q | 80, 240 |
| MCM:NDNOETLOOH | $6.9\times10^{14}$ | | Wang et al. (2017) | Q | 80, 238 |
| $C_9H_{11}N_3O_{12}$ | $1.4\times10^{9}$ | | Wang et al. (2017) | Q | 80, 239 |
| DLTVFZDBANLVJD-UHFFFAOYSA-N | $1.5\times10^{3}$ | | Wang et al. (2017) | Q | 80, 240 |
| MCM:NDNPBZLOOH | $5.1\times10^{14}$ | | Wang et al. (2017) | Q | 80, 238 |
| $C_9H_{11}N_3O_{12}$ | $6.8\times10^{8}$ | | Wang et al. (2017) | Q | 80, 239 |
| JBFCRLZRPRJLGG-UHFFFAOYSA-N | $3.6\times10^{2}$ | | Wang et al. (2017) | Q | 80, 240 |
| MCM:NDNPETLOOH | $3.8\times10^{14}$ | | Wang et al. (2017) | Q | 80, 238 |
| $C_9H_{11}N_3O_{12}$ | $6.8\times10^{8}$ | | Wang et al. (2017) | Q | 80, 239 |
| HIQGLTMZAYSJOX-UHFFFAOYSA-N | $1.7\times10^{2}$ | | Wang et al. (2017) | Q | 80, 240 |
| MCM:NDNT123OOH | $4.8\times10^{14}$ | | Wang et al. (2017) | Q | 80, 238 |
| $C_9H_{11}N_3O_{12}$ | $1.1\times10^{9}$ | | Wang et al. (2017) | Q | 80, 239 |
| UGLBOUATFGGZQD-UHFFFAOYSA-N | $1.7\times10^{3}$ | | Wang et al. (2017) | Q | 80, 240 |
| MCM:NDNT124OOH | $2.6\times10^{14}$ | | Wang et al. (2017) | Q | 80, 238 |
| $C_9H_{11}N_3O_{12}$ | $5.5\times10^{8}$ | | Wang et al. (2017) | Q | 80, 239 |
| JFYUNHMAYFMSCF-UHFFFAOYSA-N | $2.5\times10^{2}$ | | Wang et al. (2017) | Q | 80, 240 |
| MCM:NIPBNZOLOH | $7.3\times10^{8}$ | | Wang et al. (2017) | Q | 80, 238 |
| $C_9H_{13}NO_7$ | $2.1\times10^{6}$ | | Wang et al. (2017) | Q | 80, 239 |
| IAYADIRZIVDBNK-UHFFFAOYSA-N | $6.0\times10^{5}$ | | Wang et al. (2017) | Q | 80, 240 |
| MCM:NIPBZOLOOH | $1.9\times10^{10}$ | | Wang et al. (2017) | Q | 80, 238 |
| $C_9H_{13}NO_8$ | $1.2\times10^{6}$ | | Wang et al. (2017) | Q | 80, 239 |
| UJRLUYDBOZESDX-UHFFFAOYSA-N | $2.1\times10^{4}$ | | Wang et al. (2017) | Q | 80, 240 |
| MCM:NMETOLOH | $4.4\times10^{8}$ | | Wang et al. (2017) | Q | 80, 238 |
| $C_9H_{13}NO_7$ | $9.8\times10^{5}$ | | Wang et al. (2017) | Q | 80, 239 |
| JUUDFVZFULKTPV-UHFFFAOYSA-N | $1.3\times10^{4}$ | | Wang et al. (2017) | Q | 80, 240 |
| MCM:NMETOLOOH | $1.1\times10^{10}$ | | Wang et al. (2017) | Q | 80, 238 |
| $C_9H_{13}NO_8$ | $5.5\times10^{5}$ | | Wang et al. (2017) | Q | 80, 239 |
| CCCPQNSHVNXETJ-UHFFFAOYSA-N | $1.1\times10^{4}$ | | Wang et al. (2017) | Q | 80, 240 |
| MCM:NOETOLOH | $6.5\times10^{8}$ | | Wang et al. (2017) | Q | 80, 238 |
| $C_9H_{13}NO_7$ | $2.9\times10^{6}$ | | Wang et al. (2017) | Q | 80, 239 |
| RRTVORKFFCXYCH-UHFFFAOYSA-N | $4.1\times10^{3}$ | | Wang et al. (2017) | Q | 80, 240 |
| MCM:NOETOLOOH | $1.7\times10^{10}$ | | Wang et al. (2017) | Q | 80, 238 |
| $C_9H_{13}NO_8$ | $1.7\times10^{6}$ | | Wang et al. (2017) | Q | 80, 239 |
| CGRZNCSNBNPNBR-UHFFFAOYSA-N | $2.5\times10^{3}$ | | Wang et al. (2017) | Q | 80, 240 |
| MCM:NPBNZOLOH | $7.1\times10^{8}$ | | Wang et al. (2017) | Q | 80, 238 |
| $C_9H_{13}NO_7$ | $1.6\times10^{6}$ | | Wang et al. (2017) | Q | 80, 239 |
| PSGNFPQCVGIEDF-UHFFFAOYSA-N | $2.9\times10^{5}$ | | Wang et al. (2017) | Q | 80, 240 |





Table A4.6: Nitrates ($RONO_2$) (...continued)

| Substance Formula (Trivial Name) [CAS Registry Number] InChIKey | $H_s^{cp}$ (at $T^\ominus$) $\left[\dfrac{\mathrm{mol}}{\mathrm{m}^3\,\mathrm{Pa}}\right]$ | $\dfrac{\mathrm{d}\ln H_s^{cp}}{\mathrm{d}(1/T)}$ [K] | Reference | Type | Note |
|---|---|---|---|---|---|
| MCM:NPBNZOLOOH | $1.9\times10^{10}$ | | Wang et al. (2017) | Q | 80, 238 |
| $C_9H_{13}NO_8$ | $9.8\times10^{5}$ | | Wang et al. (2017) | Q | 80, 239 |
| SNFHXCYEBCZPSO-UHFFFAOYSA-N | $2.2\times10^{4}$ | | Wang et al. (2017) | Q | 80, 240 |
| MCM:NPETOLOH | $5.3\times10^{8}$ | | Wang et al. (2017) | Q | 80, 238 |
| $C_9H_{13}NO_7$ | $1.8\times10^{6}$ | | Wang et al. (2017) | Q | 80, 239 |
| ILVSHXKPUYWBMX-UHFFFAOYSA-N | $2.3\times10^{3}$ | | Wang et al. (2017) | Q | 80, 240 |
| MCM:NPETOLOOH | $1.4\times10^{10}$ | | Wang et al. (2017) | Q | 80, 238 |
| $C_9H_{13}NO_8$ | $8.9\times10^{5}$ | | Wang et al. (2017) | Q | 80, 239 |
| BKSLAYLCQMXSES-UHFFFAOYSA-N | $1.3\times10^{4}$ | | Wang et al. (2017) | Q | 80, 240 |
| MCM:NTM123LOOH | $1.1\times10^{10}$ | | Wang et al. (2017) | Q | 80, 238 |
| $C_9H_{13}NO_8$ | $9.1\times10^{5}$ | | Wang et al. (2017) | Q | 80, 239 |
| BTWUGVBELCWISQ-UHFFFAOYSA-N | $5.1\times10^{3}$ | | Wang et al. (2017) | Q | 80, 240 |
| MCM:NTM123OLOH | $4.4\times10^{8}$ | | Wang et al. (2017) | Q | 80, 238 |
| $C_9H_{13}NO_7$ | $1.6\times10^{6}$ | | Wang et al. (2017) | Q | 80, 239 |
| DXGFZPSQDYIOOU-UHFFFAOYSA-N | $7.4\times10^{3}$ | | Wang et al. (2017) | Q | 80, 240 |
| MCM:NTM124LOOH | $1.1\times10^{10}$ | | Wang et al. (2017) | Q | 80, 238 |
| $C_9H_{13}NO_8$ | $1.3\times10^{6}$ | | Wang et al. (2017) | Q | 80, 239 |
| JDMNMHRINWGUFB-UHFFFAOYSA-N | $3.4\times10^{3}$ | | Wang et al. (2017) | Q | 80, 240 |
| MCM:NTM124OLOH | $4.4\times10^{8}$ | | Wang et al. (2017) | Q | 80, 238 |
| $C_9H_{13}NO_7$ | $2.3\times10^{6}$ | | Wang et al. (2017) | Q | 80, 239 |
| RASHBSOBGFFOLP-UHFFFAOYSA-N | $2.0\times10^{3}$ | | Wang et al. (2017) | Q | 80, 240 |
| MCM:NTM135LOOH | $9.6\times10^{9}$ | | Wang et al. (2017) | Q | 80, 238 |
| $C_9H_{13}NO_8$ | $5.1\times10^{5}$ | | Wang et al. (2017) | Q | 80, 239 |
| AUGMYCIDZMPUKI-UHFFFAOYSA-N | $3.5\times10^{3}$ | | Wang et al. (2017) | Q | 80, 240 |
| MCM:NTM135OLOH | $3.6\times10^{8}$ | | Wang et al. (2017) | Q | 80, 238 |
| $C_9H_{13}NO_7$ | $1.1\times10^{6}$ | | Wang et al. (2017) | Q | 80, 239 |
| RAAKFDOLNJCXRG-UHFFFAOYSA-N | $2.7\times10^{3}$ | | Wang et al. (2017) | Q | 80, 240 |
| MCM:OETLBIPNO3 | $2.6\times10^{5}$ | | Wang et al. (2017) | Q | 80, 238 |
| $C_9H_{13}NO_6$ | $6.2\times10^{3}$ | | Wang et al. (2017) | Q | 80, 239 |
| FQBYEMVYEQQXMF-UHFFFAOYSA-N | $6.6\times10^{4}$ | | Wang et al. (2017) | Q | 80, 240 |
| MCM:OTNNCATOOH | $8.9\times10^{15}$ | | Wang et al. (2017) | Q | 80, 238 |
| $C_9H_{12}N_2O_{11}$ | $2.1\times10^{8}$ | | Wang et al. (2017) | Q | 80, 239 |
| QVFSJCAPJVJPQJ-UHFFFAOYSA-N | $1.3\times10^{4}$ | | Wang et al. (2017) | Q | 80, 240 |
| MCM:PBZBPERNO3 | $4.2\times10^{5}$ | | Wang et al. (2017) | Q | 80, 238 |
| $C_9H_{13}NO_6$ | $1.1\times10^{4}$ | | Wang et al. (2017) | Q | 80, 239 |
| XWXMYNZXCPDSAQ-UHFFFAOYSA-N | $2.1\times10^{2}$ | | Wang et al. (2017) | Q | 80, 240 |
| MCM:PETLBIPNO3 | $2.6\times10^{5}$ | | Wang et al. (2017) | Q | 80, 238 |
| $C_9H_{13}NO_6$ | $6.2\times10^{3}$ | | Wang et al. (2017) | Q | 80, 239 |
| SOIPWHDZHXSJJN-UHFFFAOYSA-N | $1.4\times10^{2}$ | | Wang et al. (2017) | Q | 80, 240 |





Table A4.6: Nitrates ($RONO_2$) (...continued)

| Substance Formula (Trivial Name) [CAS Registry Number] InChIKey | $H_s^{cp}$ (at $T^\ominus$) $\left[\dfrac{\mathrm{mol}}{\mathrm{m^3\,Pa}}\right]$ | $\dfrac{\mathrm{d}\ln H_s^{cp}}{\mathrm{d}(1/T)}$ [K] | Reference | Type | Note |
|---|---|---|---|---|---|
| MCM:PNNCATCOOH | $6.9\times10^{15}$ | | Wang et al. (2017) | Q | 80, 238 |
| $C_9H_{12}N_2O_{11}$ | $2.5\times10^{9}$ | | Wang et al. (2017) | Q | 80, 239 |
| VZHROOMDTDJZFR-UHFFFAOYSA-N | $4.3\times10^{4}$ | | Wang et al. (2017) | Q | 80, 240 |
| MCM:PTNNCATOOH | $4.8\times10^{15}$ | | Wang et al. (2017) | Q | 80, 238 |
| $C_9H_{12}N_2O_{11}$ | $1.2\times10^{8}$ | | Wang et al. (2017) | Q | 80, 239 |
| DSPOEZNZVSBNSI-UHFFFAOYSA-N | $1.8\times10^{4}$ | | Wang et al. (2017) | Q | 80, 240 |
| MCM:T123NCTOOH | $9.3\times10^{16}$ | | Wang et al. (2017) | Q | 80, 238 |
| $C_9H_{13}NO_9$ | $3.4\times10^{10}$ | | Wang et al. (2017) | Q | 80, 239 |
| QTWHDHJHZAXHRP-UHFFFAOYSA-N | $2.6\times10^{6}$ | | Wang et al. (2017) | Q | 80, 240 |
| MCM:T123NNCOOH | $3.0\times10^{15}$ | | Wang et al. (2017) | Q | 80, 238 |
| $C_9H_{12}N_2O_{11}$ | $8.9\times10^{8}$ | | Wang et al. (2017) | Q | 80, 239 |
| DNVSLQYQQCSWFT-UHFFFAOYSA-N | $1.4\times10^{5}$ | | Wang et al. (2017) | Q | 80, 240 |
| MCM:T124NCTOOH | $1.7\times10^{17}$ | | Wang et al. (2017) | Q | 80, 238 |
| $C_9H_{13}NO_9$ | $3.4\times10^{9}$ | | Wang et al. (2017) | Q | 80, 239 |
| TXEYLBVXEVQRAA-UHFFFAOYSA-N | $6.9\times10^{6}$ | | Wang et al. (2017) | Q | 80, 240 |
| MCM:T124NNCOOH | $3.0\times10^{15}$ | | Wang et al. (2017) | Q | 80, 238 |
| $C_9H_{12}N_2O_{11}$ | $1.0\times10^{8}$ | | Wang et al. (2017) | Q | 80, 239 |
| ATYVZJLHFNLHKS-UHFFFAOYSA-N | $7.8\times10^{4}$ | | Wang et al. (2017) | Q | 80, 240 |
| MCM:TM123BPNO3 | $1.8\times10^{5}$ | | Wang et al. (2017) | Q | 80, 238 |
| $C_9H_{13}NO_6$ | $4.9\times10^{3}$ | | Wang et al. (2017) | Q | 80, 239 |
| RQYOJMSOWIZBID-UHFFFAOYSA-N | $2.2\times10^{1}$ | | Wang et al. (2017) | Q | 80, 240 |
| MCM:TM124BPNO3 | $1.8\times10^{5}$ | | Wang et al. (2017) | Q | 80, 238 |
| $C_9H_{13}NO_6$ | $3.8\times10^{3}$ | | Wang et al. (2017) | Q | 80, 239 |
| KUPKWWJOBRXSMW-UHFFFAOYSA-N | $9.1\times10^{1}$ | | Wang et al. (2017) | Q | 80, 240 |
| MCM:TM135BPNO3 | $2.1\times10^{5}$ | | Wang et al. (2017) | Q | 80, 238 |
| $C_9H_{13}NO_6$ | $6.2\times10^{3}$ | | Wang et al. (2017) | Q | 80, 239 |
| VYQMVFHQGVVJTP-UHFFFAOYSA-N | $1.1\times10^{4}$ | | Wang et al. (2017) | Q | 80, 240 |
| MCM:APINANO3 | $6.9\times10^{1}$ | | Wang et al. (2017) | Q | 80, 238 |
| $C_{10}H_{17}NO_4$ | $7.3\times10^{1}$ | | Wang et al. (2017) | Q | 80, 239 |
| YITBNQHGOKCTCE-UHFFFAOYSA-N | $1.2$ | | Wang et al. (2017) | Q | 80, 240 |
| MCM:APINBNO3 | $6.9\times10^{1}$ | | Wang et al. (2017) | Q | 80, 238 |
| $C_{10}H_{17}NO_4$ | $9.3\times10^{1}$ | | Wang et al. (2017) | Q | 80, 239 |
| AFJBVFUZFQZZRG-UHFFFAOYSA-N | $2.9$ | | Wang et al. (2017) | Q | 80, 240 |
| MCM:APINCNO3 | $7.6\times10^{1}$ | | Wang et al. (2017) | Q | 80, 238 |
| $C_{10}H_{17}NO_4$ | $9.3\times10^{2}$ | | Wang et al. (2017) | Q | 80, 239 |
| IHZLDCJLQGWYSD-UHFFFAOYSA-N | $4.8\times10^{1}$ | | Wang et al. (2017) | Q | 80, 240 |
| MCM:BPINANO3 | $6.2\times10^{1}$ | | Wang et al. (2017) | Q | 80, 238 |
| $C_{10}H_{17}NO_4$ | $9.6\times10^{1}$ | | Wang et al. (2017) | Q | 80, 239 |
| LIKORQJDVXFFRR-UHFFFAOYSA-N | $5.9$ | | Wang et al. (2017) | Q | 80, 240 |





Table A4.6: Nitrates ($RONO_2$) (...continued)

| Substance<br>Formula<br>(Trivial Name)<br>[CAS Registry Number]<br>InChIKey | $H_s^{cp}$ (at $T^\ominus$) $\left[\dfrac{\mathrm{mol}}{\mathrm{m^3\,Pa}}\right]$ | $\dfrac{\mathrm{d}\ln H_s^{cp}}{\mathrm{d}(1/T)}$ [K] | Reference | Type | Note |
|---|---|---|---|---|---|
| MCM:BPINBNO3<br>$C_{10}H_{17}NO_4$<br>PIICVRQEXAAMFO-UHFFFAOYSA-N | $6.2\times10^1$<br>$1.5\times10^2$<br>$2.7$ | | Wang et al. (2017)<br>Wang et al. (2017)<br>Wang et al. (2017) | Q<br>Q<br>Q | 80, 238<br>80, 239<br>80, 240 |
| MCM:BPINCNO3<br>$C_{10}H_{17}NO_4$<br>IVSZOCZGKNSHLW-UHFFFAOYSA-N | $6.6\times10^1$<br>$1.7\times10^3$<br>$1.7\times10^2$ | | Wang et al. (2017)<br>Wang et al. (2017)<br>Wang et al. (2017) | Q<br>Q<br>Q | 80, 238<br>80, 239<br>80, 240 |
| MCM:C10PAN1<br>$C_{10}H_{19}NO_6$<br>PVQDOCOVRZVCAL-UHFFFAOYSA-N | $3.4\times10^3$<br>$6.0\times10^3$<br>$6.3\times10^{-2}$ | | Wang et al. (2017)<br>Wang et al. (2017)<br>Wang et al. (2017) | Q<br>Q<br>Q | 80, 238<br>80, 239<br>80, 240 |
| MCM:C918PAN<br>$C_{10}H_{15}NO_6$<br>GIWCNEZLGDWGNI-UHFFFAOYSA-N | $8.7\times10^2$<br>$1.1\times10^4$<br>$6.5\times10^{-1}$ | | Wang et al. (2017)<br>Wang et al. (2017)<br>Wang et al. (2017) | Q<br>Q<br>Q | 80, 238<br>80, 239<br>80, 240 |
| MCM:DMEBIPNO3<br>$C_{10}H_{15}NO_6$<br>LWYZBKFMLLCHQU-UHFFFAOYSA-N | $1.7\times10^5$<br>$4.2\times10^3$<br>$7.8\times10^1$ | | Wang et al. (2017)<br>Wang et al. (2017)<br>Wang et al. (2017) | Q<br>Q<br>Q | 80, 238<br>80, 239<br>80, 240 |
| MCM:HO3C106NO3<br>$C_{10}H_{21}NO_4$<br>LRSOGLAVONYUMB-UHFFFAOYSA-N | $1.0\times10^1$<br>$7.6\times10^1$<br>$1.0$ | | Wang et al. (2017)<br>Wang et al. (2017)<br>Wang et al. (2017) | Q<br>Q<br>Q | 80, 238<br>80, 239<br>80, 240 |
| MCM:LIMANO3<br>$C_{10}H_{17}NO_4$<br>LZDKYYHMAURBIK-UHFFFAOYSA-N | $8.4\times10^1$<br>$6.5\times10^1$<br>$1.8\times10^2$<br>$2.8$ | 12000 | Wieser et al. (2023)<br>Wang et al. (2017)<br>Wang et al. (2017)<br>Wang et al. (2017) | Q<br>Q<br>Q<br>Q | 437<br>80, 238<br>80, 239<br>80, 240 |
| MCM:LIMBNO3<br>$C_{10}H_{17}NO_4$<br>WIRXFNWUDJEZSG-UHFFFAOYSA-N | $6.5\times10^1$<br>$2.3\times10^2$<br>$5.5$ | | Wang et al. (2017)<br>Wang et al. (2017)<br>Wang et al. (2017) | Q<br>Q<br>Q | 80, 238<br>80, 239<br>80, 240 |
| MCM:LIMCNO3<br>$C_{10}H_{17}NO_4$<br>ZELLOEPLERCREX-UHFFFAOYSA-N | $8.6\times10^1$<br>$8.1\times10^1$<br>$6.2\times10^2$<br>$8.1$ | 19000 | Wieser et al. (2023)<br>Wang et al. (2017)<br>Wang et al. (2017)<br>Wang et al. (2017) | Q<br>Q<br>Q<br>Q | 437<br>80, 238<br>80, 239<br>80, 240 |
| MCM:NDMEPHOLOH<br>$C_{10}H_{15}NO_7$<br>XEZYIWLRTFENIO-UHFFFAOYSA-N | $3.0\times10^8$<br>$7.4\times10^5$<br>$1.6\times10^3$ | | Wang et al. (2017)<br>Wang et al. (2017)<br>Wang et al. (2017) | Q<br>Q<br>Q | 80, 238<br>80, 239<br>80, 240 |
| MCM:NDMEPLOOH<br>$C_{10}H_{15}NO_8$<br>CUWVWZNCHDONFF-UHFFFAOYSA-N | $7.4\times10^9$<br>$3.4\times10^5$<br>$7.6\times10^2$ | | Wang et al. (2017)<br>Wang et al. (2017)<br>Wang et al. (2017) | Q<br>Q<br>Q | 80, 238<br>80, 239<br>80, 240 |
| MCM:DETLBIPNO3<br>$C_{11}H_{17}NO_6$<br>HUKONYHKUULPQM-UHFFFAOYSA-N | $1.6\times10^5$<br>$3.1\times10^3$<br>$4.2\times10^1$ | | Wang et al. (2017)<br>Wang et al. (2017)<br>Wang et al. (2017) | Q<br>Q<br>Q | 80, 238<br>80, 239<br>80, 240 |
| MCM:HO3C116NO3<br>$C_{11}H_{23}NO_4$<br>ZSRHOFPLHKNYIP-UHFFFAOYSA-N | $8.0$<br>$6.2\times10^1$<br>$9.6\times10^{-1}$ | | Wang et al. (2017)<br>Wang et al. (2017)<br>Wang et al. (2017) | Q<br>Q<br>Q | 80, 238<br>80, 239<br>80, 240 |





Table A4.6: Nitrates ($RONO_2$) (...continued)

| Substance Formula (Trivial Name) [CAS Registry Number] InChIKey | $H_s^{cp}$ (at $T^\ominus$) $\left[\dfrac{\text{mol}}{\text{m}^3\,\text{Pa}}\right]$ | $\dfrac{\mathrm{d}\ln H_s^{cp}}{\mathrm{d}(1/T)}$ [K] | Reference | Type | Note |
|---|---|---|---|---|---|
| MCM:NDEMPHOLOH $C_{11}H_{17}NO_7$ SFAWLNJVZQJDAW-UHFFFAOYSA-N | $2.6\times10^8$ $4.7\times10^5$ $1.1\times10^3$ | | Wang et al. (2017) Wang et al. (2017) Wang et al. (2017) | Q Q Q | 80, 238 80, 239 80, 240 |
| MCM:NDEMPLOOH $C_{11}H_{17}NO_8$ KQXYYOILDOZYPH-UHFFFAOYSA-N | $6.2\times10^9$ $2.4\times10^5$ $1.5\times10^3$ | | Wang et al. (2017) Wang et al. (2017) Wang et al. (2017) | Q Q Q | 80, 238 80, 239 80, 240 |
| MCM:HO3C126NO3 $C_{12}H_{25}NO_4$ NBVYOETUGSGBPE-UHFFFAOYSA-N | $6.5$ $4.8\times10^1$ $1.1$ | | Wang et al. (2017) Wang et al. (2017) Wang et al. (2017) | Q Q Q | 80, 238 80, 239 80, 240 |
| MCM:BCANO3 $C_{15}H_{25}NO_4$ DXJZLWPVAKABFC-UHFFFAOYSA-N | $6.0\times10^1$ $1.1\times10^3$ $2.0\times10^1$ | | Wang et al. (2017) Wang et al. (2017) Wang et al. (2017) | Q Q Q | 80, 238 80, 239 80, 240 |
| MCM:BCBNO3 $C_{15}H_{25}NO_4$ GGDFBGYMWXIISJ-UHFFFAOYSA-N | $6.0\times10^1$ $1.7\times10^3$ $2.1\times10^1$ | | Wang et al. (2017) Wang et al. (2017) Wang et al. (2017) | Q Q Q | 80, 238 80, 239 80, 240 |
| MCM:BCCNO3 $C_{15}H_{25}NO_4$ AJHNYRDSMBHFTD-UHFFFAOYSA-N | $8.5\times10^1$ $2.0\times10^3$ $1.2\times10^1$ | | Wang et al. (2017) Wang et al. (2017) Wang et al. (2017) | Q Q Q | 80, 238 80, 239 80, 240 |
| MCM:GLYPAN $C_2HNO_6$ WWNQODGUPABTBB-UHFFFAOYSA-N | $4.5\times10^3$ $1.4\times10^3$ $4.2\times10^{-4}$ | | Wang et al. (2017) Wang et al. (2017) Wang et al. (2017) | Q Q Q | 80, 238 80, 239 80, 240 |
| MCM:NO3CH2CHO $C_2H_3NO_4$ ABUBKAMLUVOXSP-UHFFFAOYSA-N | $1.1\times10^1$ $9.3\times10^1$ $1.4\times10^{-1}$ | | Wang et al. (2017) Wang et al. (2017) Wang et al. (2017) | Q Q Q | 80, 238 80, 239 80, 240 |
| MCM:C3PAN2 $C_3H_3NO_6$ JCWPFKBPQPIIIA-UHFFFAOYSA-N | $3.6\times10^3$ $4.0\times10^3$ $5.9\times10^{-2}$ | | Wang et al. (2017) Wang et al. (2017) Wang et al. (2017) | Q Q Q | 80, 238 80, 239 80, 240 |
| MCM:CHOPRNO3 $C_3H_5NO_4$ IIFXHQMWCHWMFS-UHFFFAOYSA-N | $1.0\times10^1$ $3.9\times10^1$ $3.3\times10^{-2}$ | | Wang et al. (2017) Wang et al. (2017) Wang et al. (2017) | Q Q Q | 80, 238 80, 239 80, 240 |
| MCM:BUTAL2NO3 $C_4H_7NO_4$ JTMAFXAEAWTSPM-UHFFFAOYSA-N | $8.1$ $1.9\times10^1$ $1.9\times10^{-2}$ | | Wang et al. (2017) Wang et al. (2017) Wang et al. (2017) | Q Q Q | 80, 238 80, 239 80, 240 |
| MCM:BUTALNO3 $C_4H_7NO_4$ YGQOBBYMNDZOFN-UHFFFAOYSA-N | $8.1$ $4.6\times10^1$ $1.6\times10^{-1}$ | | Wang et al. (2017) Wang et al. (2017) Wang et al. (2017) | Q Q Q | 80, 238 80, 239 80, 240 |
| MCM:C4CONO3OOH $C_4H_7NO_6$ OPYZRKWHBYWVFV-UHFFFAOYSA-N | $7.1\times10^5$ $2.1\times10^5$ $1.9\times10^1$ | | Wang et al. (2017) Wang et al. (2017) Wang et al. (2017) | Q Q Q | 80, 238 80, 239 80, 240 |





Table A4.6: Nitrates ($RONO_2$) (... continued)

| Substance Formula (Trivial Name) [CAS Registry Number] InChIKey | $H_s^{cp}$ (at $T^\ominus$) $\left[\dfrac{\text{mol}}{\text{m}^3\,\text{Pa}}\right]$ | $\dfrac{\text{d}\ln H_s^{cp}}{\text{d}(1/T)}$ [K] | Reference | Type | Note |
|---|---|---|---|---|---|
| MCM:C4NO3COOOH | $7.1\times10^5$ | | Wang et al. (2017) | Q | 80, 238 |
| $C_4H_7NO_6$ | $1.1\times10^4$ | | Wang et al. (2017) | Q | 80, 239 |
| LPNJZRUFCUXQKM-UHFFFAOYSA-N | 5.9 | | Wang et al. (2017) | Q | 80, 240 |
| MCM:CHOC2PAN | $2.8\times10^3$ | | Wang et al. (2017) | Q | 80, 238 |
| $C_4H_5NO_6$ | $9.3\times10^3$ | | Wang et al. (2017) | Q | 80, 239 |
| RNHARUXEGCOCNU-UHFFFAOYSA-N | $2.6\times10^{-1}$ | | Wang et al. (2017) | Q | 80, 240 |
| MCM:CONM2CHO | $5.1\times10^3$ | | Wang et al. (2017) | Q | 80, 238 |
| $C_4H_5NO_5$ | $1.6\times10^3$ | | Wang et al. (2017) | Q | 80, 239 |
| UXLHNUQVUCPVFF-UHFFFAOYSA-N | $4.5\times10^{-2}$ | | Wang et al. (2017) | Q | 80, 240 |
| MCM:CONM2CO3H | $5.6\times10^6$ | | Wang et al. (2017) | Q | 80, 238 |
| $C_4H_5NO_7$ | $1.0\times10^5$ | | Wang et al. (2017) | Q | 80, 239 |
| KDPSYMPAERQTCV-UHFFFAOYSA-N | 1.6 | | Wang et al. (2017) | Q | 80, 240 |
| MCM:CONM2PAN | $2.6\times10^5$ | | Wang et al. (2017) | Q | 80, 238 |
| $C_4H_4N_2O_9$ | $4.5\times10^4$ | | Wang et al. (2017) | Q | 80, 239 |
| PRUSZGKEVUCSBM-UHFFFAOYSA-N | $2.8\times10^{-4}$ | | Wang et al. (2017) | Q | 80, 240 |
| MCM:IBUALANO3 | 8.1 | | Wang et al. (2017) | Q | 80, 238 |
| $C_4H_7NO_4$ | $6.0\times10^1$ | | Wang et al. (2017) | Q | 80, 239 |
| BBEYVRKCPRBWGK-UHFFFAOYSA-N | $2.0\times10^{-1}$ | | Wang et al. (2017) | Q | 80, 240 |
| MCM:IBUDIALPAN | $3.3\times10^3$ | | Wang et al. (2017) | Q | 80, 238 |
| $C_4H_5NO_6$ | $1.4\times10^3$ | | Wang et al. (2017) | Q | 80, 239 |
| ANKFJODHGNELSW-UHFFFAOYSA-N | $6.0\times10^{-3}$ | | Wang et al. (2017) | Q | 80, 240 |
| MCM:MALDIALPAN | $1.1\times10^4$ | | Wang et al. (2017) | Q | 80, 238 |
| $C_4H_3NO_6$ | $2.3\times10^4$ | | Wang et al. (2017) | Q | 80, 239 |
| JVYQADFHFGBZRY-UHFFFAOYSA-N | $2.2\times10^{-2}$ | | Wang et al. (2017) | Q | 80, 240 |
| MCM:MPRBNO3CHO | 5.6 | | Wang et al. (2017) | Q | 80, 238 |
| $C_4H_7NO_4$ | 7.8 | | Wang et al. (2017) | Q | 80, 239 |
| ZFRSNPXDNOXJAQ-UHFFFAOYSA-N | $1.3\times10^{-2}$ | | Wang et al. (2017) | Q | 80, 240 |
| MCM:NC3CHO | $3.3\times10^1$ | | Wang et al. (2017) | Q | 80, 238 |
| $C_4H_5NO_4$ | $1.1\times10^2$ | | Wang et al. (2017) | Q | 80, 239 |
| IRCIUAVPVOGPAB-UHFFFAOYSA-N | 3.7 | | Wang et al. (2017) | Q | 80, 240 |
| MCM:C3M3CHONO3 | 4.7 | | Wang et al. (2017) | Q | 80, 238 |
| $C_5H_9NO_4$ | $1.6\times10^1$ | | Wang et al. (2017) | Q | 80, 239 |
| FSONNVXXYIKSKL-UHFFFAOYSA-N | $1.7\times10^{-1}$ | | Wang et al. (2017) | Q | 80, 240 |
| MCM:C3MCODBPAN | $7.6\times10^3$ | | Wang et al. (2017) | Q | 80, 238 |
| $C_5H_5NO_6$ | $2.2\times10^4$ | | Wang et al. (2017) | Q | 80, 239 |
| ABUKHSDGKGHQTM-UHFFFAOYSA-N | $6.8\times10^{-3}$ | | Wang et al. (2017) | Q | 80, 240 |
| MCM:C3MNO3CHO | 7.6 | | Wang et al. (2017) | Q | 80, 238 |
| $C_5H_9NO_4$ | $1.4\times10^1$ | | Wang et al. (2017) | Q | 80, 239 |
| IIGIHDSEOSVDKL-UHFFFAOYSA-N | $1.9\times10^{-2}$ | | Wang et al. (2017) | Q | 80, 240 |



Table A4.6: Nitrates ($RONO_2$) (...continued)

| Substance Formula (Trivial Name) [CAS Registry Number] InChIKey | $H_s^{cp}$ (at $T^\ominus$) $\left[\dfrac{\text{mol}}{\text{m}^3\,\text{Pa}}\right]$ | $\dfrac{\mathrm{d}\ln H_s^{cp}}{\mathrm{d}(1/T)}$ [K] | Reference | Type | Note |
|---|---|---|---|---|---|
| MCM:C4CHOBNO3 | 6.8 | | Wang et al. (2017) | Q | 80, 238 |
| $C_5H_9NO_4$ | $2.5\times10^1$ | | Wang et al. (2017) | Q | 80, 239 |
| IRTPVYBICUATCF-UHFFFAOYSA-N | $9.6\times10^{-2}$ | | Wang et al. (2017) | Q | 80, 240 |
| MCM:C4NO3CHO | 4.7 | | Wang et al. (2017) | Q | 80, 238 |
| $C_5H_9NO_4$ | 5.0 | | Wang et al. (2017) | Q | 80, 239 |
| WGTULXFSTHCAFV-UHFFFAOYSA-N | $1.0\times10^{-2}$ | | Wang et al. (2017) | Q | 80, 240 |
| MCM:C514NO3 | $6.0\times10^3$ | | Wang et al. (2017) | Q | 80, 238 |
| $C_5H_7NO_5$ | $3.6\times10^4$ | | Wang et al. (2017) | Q | 80, 239 |
| CEGGSEAULUFJEH-UHFFFAOYSA-N | $2.8\times10^1$ | | Wang et al. (2017) | Q | 80, 240 |
| MCM:C52NO31CO | 6.8 | | Wang et al. (2017) | Q | 80, 238 |
| $C_5H_9NO_4$ | $1.2\times10^1$ | | Wang et al. (2017) | Q | 80, 239 |
| OIPNOUWHATXHHM-UHFFFAOYSA-N | $1.6\times10^{-2}$ | | Wang et al. (2017) | Q | 80, 240 |
| MCM:CO1M22PAN | $1.8\times10^3$ | | Wang et al. (2017) | Q | 80, 238 |
| $C_5H_7NO_6$ | $6.0\times10^2$ | | Wang et al. (2017) | Q | 80, 239 |
| JNBWLVYXFYINSW-UHFFFAOYSA-N | $4.4\times10^{-3}$ | | Wang et al. (2017) | Q | 80, 240 |
| MCM:MC3CODBPAN | $7.6\times10^3$ | | Wang et al. (2017) | Q | 80, 238 |
| $C_5H_5NO_6$ | $1.9\times10^4$ | | Wang et al. (2017) | Q | 80, 239 |
| AENWJVOXGCGHKR-UHFFFAOYSA-N | $4.1\times10^{-2}$ | | Wang et al. (2017) | Q | 80, 240 |
| MCM:NC4CHO | $2.2\times10^1$ | | Wang et al. (2017) | Q | 80, 238 |
| $C_5H_7NO_4$ | $9.8\times10^1$ | | Wang et al. (2017) | Q | 80, 239 |
| FHQODWHGFJXLCS-UHFFFAOYSA-N | 6.3 | | Wang et al. (2017) | Q | 80, 240 |
| MCM:C522PAN | $4.2\times10^3$ | | Wang et al. (2017) | Q | 80, 238 |
| $C_6H_7NO_6$ | $4.6\times10^3$ | | Wang et al. (2017) | Q | 80, 239 |
| BLDVCFQHUDZGIY-UHFFFAOYSA-N | $9.8\times10^{-2}$ | | Wang et al. (2017) | Q | 80, 240 |
| MCM:C65NO36CHO | 5.9 | | Wang et al. (2017) | Q | 80, 238 |
| $C_6H_{11}NO_4$ | 8.9 | | Wang et al. (2017) | Q | 80, 239 |
| AWOTULOVYXZRSX-UHFFFAOYSA-N | $1.4\times10^{-2}$ | | Wang et al. (2017) | Q | 80, 240 |
| MCM:CHOC4PAN | $2.0\times10^3$ | | Wang et al. (2017) | Q | 80, 238 |
| $C_6H_9NO_6$ | $3.6\times10^3$ | | Wang et al. (2017) | Q | 80, 239 |
| LKJHECXHBJSEQK-UHFFFAOYSA-N | $3.1\times10^{-1}$ | | Wang et al. (2017) | Q | 80, 240 |
| MCM:CO1C6NO3 | 5.3 | | Wang et al. (2017) | Q | 80, 238 |
| $C_6H_{11}NO_4$ | $2.9\times10^1$ | | Wang et al. (2017) | Q | 80, 239 |
| BSOUFWBHIRXMPX-UHFFFAOYSA-N | 1.8 | | Wang et al. (2017) | Q | 80, 240 |
| MCM:C615PAN | $1.2\times10^6$ | | Wang et al. (2017) | Q | 80, 238 |
| $C_7H_9NO_7$ | $2.3\times10^5$ | | Wang et al. (2017) | Q | 80, 239 |
| QXDOZRAOCVERTF-UHFFFAOYSA-N | $1.4\times10^{-1}$ | | Wang et al. (2017) | Q | 80, 240 |
| MCM:C729NO3 | $1.1\times10^1$ | | Wang et al. (2017) | Q | 80, 238 |
| $C_7H_{11}NO_4$ | $3.2\times10^1$ | | Wang et al. (2017) | Q | 80, 239 |
| IVMGVFSKIJXXOW-UHFFFAOYSA-N | $4.7\times10^{-1}$ | | Wang et al. (2017) | Q | 80, 240 |





Table A4.6: Nitrates ($RONO_2$) (... continued)

| Substance<br>Formula<br>(Trivial Name)<br>[CAS Registry Number]<br>InChIKey | $H_s^{cp}$<br>(at $T^\ominus$)<br>$\left[\dfrac{\text{mol}}{\text{m}^3\,\text{Pa}}\right]$ | $\dfrac{\text{d}\ln H_s^{cp}}{\text{d}(1/T)}$<br><br>[K] | Reference | Type | Note |
|---|---|---|---|---|---|
| MCM:C729PAN<br>$C_8H_{11}NO_6$<br>UQYKHYVNMQUYAN-UHFFFAOYSA-N | $3.6\times10^3$<br>$1.6\times10^3$<br>$2.0\times10^{-1}$ | | Wang et al. (2017)<br>Wang et al. (2017)<br>Wang et al. (2017) | Q<br>Q<br>Q | 80, 238<br>80, 239<br>80, 240 |
| MCM:C810NO3<br>$C_8H_{13}NO_5$<br>CHBSGOHAWLCLGZ-UHFFFAOYSA-N | $2.8\times10^3$<br>$7.3\times10^3$<br>$4.3\times10^2$ | | Wang et al. (2017)<br>Wang et al. (2017)<br>Wang et al. (2017) | Q<br>Q<br>Q | 80, 238<br>80, 239<br>80, 240 |
| MCM:C822NO3<br>$C_8H_{13}NO_4$<br>JWAJNALJDDSHOA-UHFFFAOYSA-N | $2.9\times10^1$<br>8.9<br>$2.6\times10^1$<br>1.1 | 10000 | Wieser et al. (2023)<br>Wang et al. (2017)<br>Wang et al. (2017)<br>Wang et al. (2017) | Q<br>Q<br>Q<br>Q | 437<br>80, 238<br>80, 239<br>80, 240 |
| MCM:C830NO3<br>$C_8H_{13}NO_4$<br>FXBNRUYCAFEPCZ-UHFFFAOYSA-N | $1.0\times10^1$<br>$2.8\times10^1$<br>$7.6\times10^{-1}$ | | Wang et al. (2017)<br>Wang et al. (2017)<br>Wang et al. (2017) | Q<br>Q<br>Q | 80, 238<br>80, 239<br>80, 240 |
| MCM:C831NO3<br>$C_8H_{13}NO_5$<br>OVEOYJGENOZMLJ-UHFFFAOYSA-N | $2.8\times10^3$<br>$9.3\times10^3$<br>$5.3\times10^1$ | | Wang et al. (2017)<br>Wang et al. (2017)<br>Wang et al. (2017) | Q<br>Q<br>Q | 80, 238<br>80, 239<br>80, 240 |
| MCM:C89NO3<br>$C_8H_{13}NO_4$<br>JVGNHSSLWSUTIF-UHFFFAOYSA-N | $1.0\times10^1$<br>$2.8\times10^1$<br>2.0 | | Wang et al. (2017)<br>Wang et al. (2017)<br>Wang et al. (2017) | Q<br>Q<br>Q | 80, 238<br>80, 239<br>80, 240 |
| MCM:NC826OOH<br>$C_8H_{13}NO_7$<br>LIEZCAAIBRALIF-UHFFFAOYSA-N | $1.9\times10^8$<br>$5.1\times10^7$<br>$3.6\times10^4$ | | Wang et al. (2017)<br>Wang et al. (2017)<br>Wang et al. (2017) | Q<br>Q<br>Q | 80, 238<br>80, 239<br>80, 240 |
| MCM:C822PAN<br>$C_9H_{13}NO_6$<br>AEOJBADXVLGVNE-UHFFFAOYSA-N | $2.8\times10^3$<br>$1.1\times10^3$<br>$3.4\times10^{-1}$ | | Wang et al. (2017)<br>Wang et al. (2017)<br>Wang et al. (2017) | Q<br>Q<br>Q | 80, 238<br>80, 239<br>80, 240 |
| MCM:C830PAN<br>$C_9H_{13}NO_6$<br>NSAJKAFLWKCEMC-UHFFFAOYSA-N | $3.2\times10^3$<br>$2.4\times10^3$<br>$1.1\times10^{-1}$ | | Wang et al. (2017)<br>Wang et al. (2017)<br>Wang et al. (2017) | Q<br>Q<br>Q | 80, 238<br>80, 239<br>80, 240 |
| MCM:C89PAN<br>$C_9H_{13}NO_6$<br>MAIUDZACSGCMBD-UHFFFAOYSA-N | $3.2\times10^3$<br>$2.5\times10^3$<br>1.4 | | Wang et al. (2017)<br>Wang et al. (2017)<br>Wang et al. (2017) | Q<br>Q<br>Q | 80, 238<br>80, 239<br>80, 240 |
| MCM:NC91CHO<br>$C_{10}H_{15}NO_4$<br>AVQCYYCZMFYNOE-UHFFFAOYSA-N | $1.6\times10^1$<br>$1.4\times10^1$<br>$8.1\times10^{-2}$ | | Wang et al. (2017)<br>Wang et al. (2017)<br>Wang et al. (2017) | Q<br>Q<br>Q | 80, 238<br>80, 239<br>80, 240 |
| MCM:C126NO3<br>$C_{12}H_{19}NO_4$<br>LCRFSVKVQOPOBC-UHFFFAOYSA-N | $1.1\times10^1$<br>$3.4\times10^1$<br>7.3 | | Wang et al. (2017)<br>Wang et al. (2017)<br>Wang et al. (2017) | Q<br>Q<br>Q | 80, 238<br>80, 239<br>80, 240 |
| MCM:C126PAN<br>$C_{13}H_{19}NO_6$<br>TVUSLNPYCCAOKX-UHFFFAOYSA-N | $3.4\times10^3$<br>$1.8\times10^3$<br>1.4 | | Wang et al. (2017)<br>Wang et al. (2017)<br>Wang et al. (2017) | Q<br>Q<br>Q | 80, 238<br>80, 239<br>80, 240 |





Table A4.6: Nitrates ($RONO_2$) (...continued)

| Substance Formula (Trivial Name) [CAS Registry Number] InChIKey | $H_s^{cp}$ (at $T^{\ominus}$) $\left[\dfrac{\text{mol}}{\text{m}^3\,\text{Pa}}\right]$ | $\dfrac{\text{d}\ln H_s^{cp}}{\text{d}(1/T)}$ [K] | Reference | Type | Note |
|---|---|---|---|---|---|
| MCM:C136NO3 | 8.7 | | Wang et al. (2017) | Q | 80, 238 |
| $C_{13}H_{21}NO_4$ | $3.0\times10^1$ | | Wang et al. (2017) | Q | 80, 239 |
| FAQBCXSJSFDOSN-UHFFFAOYSA-N | 8.0 | | Wang et al. (2017) | Q | 80, 240 |
| MCM:NC1313OOH | $2.0\times10^8$ | | Wang et al. (2017) | Q | 80, 238 |
| $C_{13}H_{21}NO_7$ | $9.1\times10^7$ | | Wang et al. (2017) | Q | 80, 239 |
| AIWUKZRYSUWIQN-UHFFFAOYSA-N | $2.3\times10^3$ | | Wang et al. (2017) | Q | 80, 240 |
| MCM:C136PAN | $3.1\times10^3$ | | Wang et al. (2017) | Q | 80, 238 |
| $C_{14}H_{21}NO_6$ | $1.4\times10^3$ | | Wang et al. (2017) | Q | 80, 239 |
| HVYQJFOLIDVKQK-UHFFFAOYSA-N | 4.1 | | Wang et al. (2017) | Q | 80, 240 |
| MCM:C42AOH | $1.8\times10^3$ | | Wang et al. (2017) | Q | 80, 238 |
| $C_3H_5NO_5$ | $1.6\times10^4$ | | Wang et al. (2017) | Q | 80, 239 |
| CQVFGUOZVNZZAM-UHFFFAOYSA-N | 5.5 | | Wang et al. (2017) | Q | 80, 240 |
| MCM:HCOCOHPAN | $1.2\times10^5$ | | Wang et al. (2017) | Q | 80, 238 |
| $C_3H_3NO_7$ | $5.4\times10^5$ | | Wang et al. (2017) | Q | 80, 239 |
| GGQYFRDCXKQQIA-UHFFFAOYSA-N | $2.1\times10^{-1}$ | | Wang et al. (2017) | Q | 80, 240 |
| MCM:C41NO3 | $5.0\times10^6$ | | Wang et al. (2017) | Q | 80, 238 |
| $C_4H_7NO_6$ | $1.4\times10^7$ | | Wang et al. (2017) | Q | 80, 239 |
| SGYMVPWBXCYWGY-UHFFFAOYSA-N | $1.1\times10^3$ | | Wang et al. (2017) | Q | 80, 240 |
| MCM:C42OH | $1.8\times10^6$ | | Wang et al. (2017) | Q | 80, 238 |
| $C_4H_7NO_6$ | $1.6\times10^7$ | | Wang et al. (2017) | Q | 80, 239 |
| OIAGCOLJHCONRV-UHFFFAOYSA-N | $1.7\times10^2$ | | Wang et al. (2017) | Q | 80, 240 |
| MCM:C42OOH | $1.3\times10^8$ | | Wang et al. (2017) | Q | 80, 238 |
| $C_4H_7NO_7$ | $1.4\times10^7$ | | Wang et al. (2017) | Q | 80, 239 |
| KUKJNXCETLVUQA-UHFFFAOYSA-N | $1.8\times10^2$ | | Wang et al. (2017) | Q | 80, 240 |
| MCM:C4OCCOHNO3 | $1.7\times10^3$ | | Wang et al. (2017) | Q | 80, 238 |
| $C_4H_7NO_5$ | $9.8\times10^3$ | | Wang et al. (2017) | Q | 80, 239 |
| NEHFOAWUQWHZTI-UHFFFAOYSA-N | 1.6 | | Wang et al. (2017) | Q | 80, 240 |
| MCM:COCCOHNO3 | $3.3\times10^4$ | | Wang et al. (2017) | Q | 80, 238 |
| $C_4H_7NO_5$ | $3.2\times10^4$ | | Wang et al. (2017) | Q | 80, 239 |
| HMCDHWALZVCNKO-UHFFFAOYSA-N | 6.8 | | Wang et al. (2017) | Q | 80, 240 |
| MCM:COHM2PAN | $6.5\times10^4$ | | Wang et al. (2017) | Q | 80, 238 |
| $C_4H_5NO_7$ | $2.2\times10^5$ | | Wang et al. (2017) | Q | 80, 239 |
| UXUAXYPQPQSXRR-UHFFFAOYSA-N | $5.0\times10^{-2}$ | | Wang et al. (2017) | Q | 80, 240 |
| MCM:MACRNB | $1.0\times10^3$ | | Wang et al. (2017) | Q | 80, 238 |
| $C_4H_7NO_5$ | $1.3\times10^4$ | | Wang et al. (2017) | Q | 80, 239 |
| DLZDCWJOERFHAS-UHFFFAOYSA-N | 1.1 | | Wang et al. (2017) | Q | 80, 240 |
| MCM:MACRNO3 | $2.0\times10^4$ | | Wang et al. (2017) | Q | 80, 238 |
| $C_4H_7NO_5$ | $7.1\times10^3$ | | Wang et al. (2017) | Q | 80, 239 |
| ALINXPRBVMXOGH-UHFFFAOYSA-N | 3.2 | | Wang et al. (2017) | Q | 80, 240 |





Table A4.6: Nitrates ($RONO_2$) (...continued)

| Substance Formula (Trivial Name) [CAS Registry Number] InChIKey | $H_s^{cp}$ (at $T^{\ominus}$) $\left[\dfrac{mol}{m^3\,Pa}\right]$ | $\dfrac{d\ln H_s^{cp}}{d(1/T)}$ [K] | Reference | Type | Note |
|---|---|---|---|---|---|
| MCM:C4M2ALOHNO3 | $8.5\times10^5$ | | Wang et al. (2017) | Q | 80, 238 |
| $C_5H_7NO_6$ | $1.4\times10^6$ | | Wang et al. (2017) | Q | 80, 239 |
| UOGOTHSJBRKVRR-UHFFFAOYSA-N | 2.2 | | Wang et al. (2017) | Q | 80, 240 |
| MCM:C510OH | $1.0\times10^6$ | | Wang et al. (2017) | Q | 80, 238 |
| $C_5H_9NO_6$ | $1.2\times10^7$ | | Wang et al. (2017) | Q | 80, 239 |
| ZCOZZYNQPKXJEJ-UHFFFAOYSA-N | $8.3\times10^1$ | | Wang et al. (2017) | Q | 80, 240 |
| MCM:C510OOH | $7.1\times10^7$ | | Wang et al. (2017) | Q | 80, 238 |
| $C_5H_9NO_7$ | $1.4\times10^7$ | | Wang et al. (2017) | Q | 80, 239 |
| GGBWELVRYZRQMP-UHFFFAOYSA-N | $1.8\times10^3$ | | Wang et al. (2017) | Q | 80, 240 |
| MCM:C57NO3 | $2.0\times10^7$ | | Wang et al. (2017) | Q | 80, 238 |
| $C_5H_9NO_6$ | $1.3\times10^7$ | | Wang et al. (2017) | Q | 80, 239 |
| KHAJTJUOMDNQGW-UHFFFAOYSA-N | $6.8\times10^2$ | | Wang et al. (2017) | Q | 80, 240 |
| MCM:C58NO3 | $2.8\times10^6$ | | Wang et al. (2017) | Q | 80, 238 |
| $C_5H_9NO_6$ | $1.0\times10^7$ | | Wang et al. (2017) | Q | 80, 239 |
| UZUIBKBJXOKLGO-UHFFFAOYSA-N | $5.1\times10^2$ | | Wang et al. (2017) | Q | 80, 240 |
| MCM:DNC524CO | $7.1\times10^9$ | | Wang et al. (2017) | Q | 80, 238 |
| $C_5H_8N_2O_9$ | $9.6\times10^8$ | | Wang et al. (2017) | Q | 80, 239 |
| WXHJTLSKHGDJHS-UHFFFAOYSA-N | $1.9\times10^4$ | | Wang et al. (2017) | Q | 80, 240 |
| MCM:HNC524CO | $2.9\times10^9$ | | Wang et al. (2017) | Q | 80, 238 |
| $C_5H_9NO_7$ | $1.5\times10^{10}$ | | Wang et al. (2017) | Q | 80, 239 |
| UHQMTOKWUFCERL-UHFFFAOYSA-N | $5.9\times10^4$ | | Wang et al. (2017) | Q | 80, 240 |
| MCM:HPNC524CO | $4.7\times10^{12}$ | | Wang et al. (2017) | Q | 80, 238 |
| $C_5H_9NO_8$ | $1.7\times10^{10}$ | | Wang et al. (2017) | Q | 80, 239 |
| QASKBBWYYSGCPQ-UHFFFAOYSA-N | $1.9\times10^5$ | | Wang et al. (2017) | Q | 80, 240 |
| MCM:INAHCHO | $1.0\times10^6$ | | Wang et al. (2017) | Q | 80, 238 |
| $C_5H_9NO_6$ | $1.5\times10^7$ | | Wang et al. (2017) | Q | 80, 239 |
| GJDCYRXOLZPRRL-UHFFFAOYSA-N | $1.2\times10^2$ | | Wang et al. (2017) | Q | 80, 240 |
| MCM:INAHPCHO | $1.6\times10^9$ | | Wang et al. (2017) | Q | 80, 238 |
| $C_5H_9NO_7$ | $2.6\times10^7$ | | Wang et al. (2017) | Q | 80, 239 |
| WOIAKHQAPCBXNP-UHFFFAOYSA-N | $1.1\times10^2$ | | Wang et al. (2017) | Q | 80, 240 |
| MCM:INANCHO | $2.5\times10^6$ | | Wang et al. (2017) | Q | 80, 238 |
| $C_5H_8N_2O_8$ | $1.7\times10^6$ | | Wang et al. (2017) | Q | 80, 239 |
| AOWQROMNZIXQRL-UHFFFAOYSA-N | 6.3 | | Wang et al. (2017) | Q | 80, 240 |
| MCM:INB1HPCHO | $1.4\times10^9$ | | Wang et al. (2017) | Q | 80, 238 |
| $C_5H_9NO_7$ | $1.9\times10^7$ | | Wang et al. (2017) | Q | 80, 239 |
| PPUPDRIHXISUBS-UHFFFAOYSA-N | $1.7\times10^3$ | | Wang et al. (2017) | Q | 80, 240 |
| MCM:INB1NACHO | $2.0\times10^6$ | | Wang et al. (2017) | Q | 80, 238 |
| $C_5H_8N_2O_8$ | $7.4\times10^6$ | | Wang et al. (2017) | Q | 80, 239 |
| FPDRKJBIBBHIBW-UHFFFAOYSA-N | $1.4\times10^1$ | | Wang et al. (2017) | Q | 80, 240 |





Table A4.6: Nitrates ($RONO_2$) (...continued)

| Substance / Formula / (Trivial Name) / [CAS Registry Number] / InChIKey | $H_s^{cp}$ (at $T^{\ominus}$) $\left[\dfrac{\text{mol}}{\text{m}^3\,\text{Pa}}\right]$ | $\dfrac{\mathrm{d}\ln H_s^{cp}}{\mathrm{d}(1/T)}$ [K] | Reference | Type | Note |
|---|---|---|---|---|---|
| MCM:INB1NBCHO | $2.0\times10^6$ | | Wang et al. (2017) | Q | 80, 238 |
| $C_5H_8N_2O_8$ | $5.4\times10^6$ | | Wang et al. (2017) | Q | 80, 239 |
| DIJODJNNHWDXRN-UHFFFAOYSA-N | $2.7\times10^1$ | | Wang et al. (2017) | Q | 80, 240 |
| MCM:INCNCHO | $2.5\times10^6$ | | Wang et al. (2017) | Q | 80, 238 |
| $C_5H_8N_2O_8$ | $3.4\times10^6$ | | Wang et al. (2017) | Q | 80, 239 |
| QYWHAZCEYLWYSZ-UHFFFAOYSA-N | 3.1 | | Wang et al. (2017) | Q | 80, 240 |
| MCM:INDHCHO | $2.8\times10^6$ | | Wang et al. (2017) | Q | 80, 238 |
| $C_5H_9NO_6$ | $1.6\times10^7$ | | Wang et al. (2017) | Q | 80, 239 |
| LXJORHVMGZGMRK-UHFFFAOYSA-N | $2.5\times10^2$ | | Wang et al. (2017) | Q | 80, 240 |
| MCM:INDHPCHO | $1.4\times10^9$ | | Wang et al. (2017) | Q | 80, 238 |
| $C_5H_9NO_7$ | $1.3\times10^7$ | | Wang et al. (2017) | Q | 80, 239 |
| NKPKKBYOCYFEQQ-UHFFFAOYSA-N | $4.0\times10^3$ | | Wang et al. (2017) | Q | 80, 240 |
| MCM:MMALNACO3H | $9.3\times10^8$ | | Wang et al. (2017) | Q | 80, 238 |
| $C_5H_7NO_8$ | $2.0\times10^6$ | | Wang et al. (2017) | Q | 80, 239 |
| WWJYGONDDIBFHK-UHFFFAOYSA-N | $8.0\times10^2$ | | Wang et al. (2017) | Q | 80, 240 |
| MCM:MMALNAPAN | $4.3\times10^7$ | | Wang et al. (2017) | Q | 80, 238 |
| $C_5H_6N_2O_{10}$ | $6.0\times10^7$ | | Wang et al. (2017) | Q | 80, 239 |
| JNHZFZYBXYVXPX-UHFFFAOYSA-N | $2.3\times10^{-2}$ | | Wang et al. (2017) | Q | 80, 240 |
| MCM:MMALNBCO3H | $9.3\times10^8$ | | Wang et al. (2017) | Q | 80, 238 |
| $C_5H_7NO_8$ | $3.9\times10^7$ | | Wang et al. (2017) | Q | 80, 239 |
| ASZGKTQVJXPLHM-UHFFFAOYSA-N | $2.5\times10^2$ | | Wang et al. (2017) | Q | 80, 240 |
| MCM:MMALNBPAN | $4.3\times10^7$ | | Wang et al. (2017) | Q | 80, 238 |
| $C_5H_6N_2O_{10}$ | $1.3\times10^7$ | | Wang et al. (2017) | Q | 80, 239 |
| QNDDWZGVLTYMIK-UHFFFAOYSA-N | $1.1\times10^{-2}$ | | Wang et al. (2017) | Q | 80, 240 |
| MCM:NMBOBCO | $1.8\times10^4$ | | Wang et al. (2017) | Q | 80, 238 |
| $C_5H_9NO_5$ | $2.2\times10^4$ | | Wang et al. (2017) | Q | 80, 239 |
| ZWLCCOCHHFERQT-UHFFFAOYSA-N | 1.6 | | Wang et al. (2017) | Q | 80, 240 |
| MCM:C1H4C5PAN | $5.9\times10^6$ | | Wang et al. (2017) | Q | 80, 238 |
| $C_6H_9NO_7$ | $2.5\times10^7$ | | Wang et al. (2017) | Q | 80, 239 |
| XKXYOEFJYDLELH-UHFFFAOYSA-N | $1.0\times10^1$ | | Wang et al. (2017) | Q | 80, 240 |
| MCM:C623NO3 | $4.2\times10^7$ | | Wang et al. (2017) | Q | 80, 238 |
| $C_6H_{11}NO_6$ | $6.0\times10^7$ | | Wang et al. (2017) | Q | 80, 239 |
| UTKYBRKMQKBGOV-UHFFFAOYSA-N | $2.2\times10^4$ | | Wang et al. (2017) | Q | 80, 240 |
| MCM:C67NO3 | $1.5\times10^4$ | | Wang et al. (2017) | Q | 80, 238 |
| $C_6H_{11}NO_5$ | $6.9\times10^4$ | | Wang et al. (2017) | Q | 80, 239 |
| MTROCBFSNOSSFX-UHFFFAOYSA-N | $4.3\times10^1$ | | Wang et al. (2017) | Q | 80, 240 |
| MCM:C68NO3 | $1.5\times10^4$ | | Wang et al. (2017) | Q | 80, 238 |
| $C_6H_{11}NO_5$ | $6.8\times10^4$ | | Wang et al. (2017) | Q | 80, 239 |
| DFCRZDMTYDUCKH-UHFFFAOYSA-N | $1.6\times10^1$ | | Wang et al. (2017) | Q | 80, 240 |





Table A4.6: Nitrates ($RONO_2$) (. . . continued)

| Substance Formula (Trivial Name) [CAS Registry Number] InChIKey | $H_s^{cp}$ (at $T^\ominus$) $\left[\dfrac{\text{mol}}{\text{m}^3\,\text{Pa}}\right]$ | $\dfrac{\text{d}\ln H_s^{cp}}{\text{d}(1/T)}$ [K] | Reference | Type | Note |
|---|---|---|---|---|---|
| MCM:CO1H63NO3 | $1.4\times10^4$ | | Wang et al. (2017) | Q | 80, 238 |
| $C_6H_{11}NO_5$ | $6.0\times10^5$ | | Wang et al. (2017) | Q | 80, 239 |
| QWPYKWKOPQKYFO-UHFFFAOYSA-N | $1.7\times10^3$ | | Wang et al. (2017) | Q | 80, 240 |
| MCM:NC623OH | $4.2\times10^7$ | | Wang et al. (2017) | Q | 80, 238 |
| $C_6H_{11}NO_6$ | $2.7\times10^8$ | | Wang et al. (2017) | Q | 80, 239 |
| LIALSLHETRNECK-UHFFFAOYSA-N | $2.3\times10^2$ | | Wang et al. (2017) | Q | 80, 240 |
| MCM:NC623OOH | $1.0\times10^9$ | | Wang et al. (2017) | Q | 80, 238 |
| $C_6H_{11}NO_7$ | $6.0\times10^7$ | | Wang et al. (2017) | Q | 80, 239 |
| LUCHFEJMQFYPEG-UHFFFAOYSA-N | $6.9\times10^4$ | | Wang et al. (2017) | Q | 80, 240 |
| MCM:C728NO3 | $1.5\times10^8$ | 17000 | Wieser et al. (2023) | Q | 437 |
| $C_7H_{13}NO_6$ | $3.2\times10^7$ | | Wang et al. (2017) | Q | 80, 238 |
| CDBHNFBQKSTCON-UHFFFAOYSA-N | $2.0\times10^8$ | | Wang et al. (2017) | Q | 80, 239 |
| | $3.0\times10^5$ | | Wang et al. (2017) | Q | 80, 240 |
| MCM:C730NO3 | $3.5\times10^9$ | 17000 | Wieser et al. (2023) | Q | 437 |
| $C_7H_{13}NO_6$ | $3.2\times10^7$ | | Wang et al. (2017) | Q | 80, 238 |
| RVWIAVBELSCRER-UHFFFAOYSA-N | $1.4\times10^9$ | | Wang et al. (2017) | Q | 80, 239 |
| | $9.8\times10^4$ | | Wang et al. (2017) | Q | 80, 240 |
| MCM:NC728OH | $3.2\times10^7$ | | Wang et al. (2017) | Q | 80, 238 |
| $C_7H_{13}NO_6$ | $5.4\times10^8$ | | Wang et al. (2017) | Q | 80, 239 |
| RMGZPHZIJLBXBA-UHFFFAOYSA-N | $9.8\times10^4$ | | Wang et al. (2017) | Q | 80, 240 |
| MCM:NC728OOH | $7.8\times10^8$ | | Wang et al. (2017) | Q | 80, 238 |
| $C_7H_{13}NO_7$ | $2.0\times10^8$ | | Wang et al. (2017) | Q | 80, 239 |
| NMPAVXKBWDGSMK-UHFFFAOYSA-N | $2.1\times10^5$ | | Wang et al. (2017) | Q | 80, 240 |
| MCM:NC730OH | $3.2\times10^7$ | | Wang et al. (2017) | Q | 80, 238 |
| $C_7H_{13}NO_6$ | $2.4\times10^8$ | | Wang et al. (2017) | Q | 80, 239 |
| ZPFRZNAADTZJOS-UHFFFAOYSA-N | $1.5\times10^4$ | | Wang et al. (2017) | Q | 80, 240 |
| MCM:NC730OOH | $7.8\times10^8$ | | Wang et al. (2017) | Q | 80, 238 |
| $C_7H_{13}NO_7$ | $2.7\times10^9$ | | Wang et al. (2017) | Q | 80, 239 |
| LJYXGYOAXKBDIY-UHFFFAOYSA-N | $6.0\times10^4$ | | Wang et al. (2017) | Q | 80, 240 |
| MCM:C826NO3 | $9.8\times10^6$ | | Wang et al. (2017) | Q | 80, 238 |
| $C_8H_{13}NO_6$ | $4.6\times10^7$ | | Wang et al. (2017) | Q | 80, 239 |
| HYUHCBAMOFNBPC-UHFFFAOYSA-N | $5.5\times10^4$ | | Wang et al. (2017) | Q | 80, 240 |
| MCM:NC826OH | $9.8\times10^6$ | | Wang et al. (2017) | Q | 80, 238 |
| $C_8H_{13}NO_6$ | $6.2\times10^7$ | | Wang et al. (2017) | Q | 80, 239 |
| UXZRCYKYQRKKOI-UHFFFAOYSA-N | $3.5\times10^3$ | | Wang et al. (2017) | Q | 80, 240 |
| MCM:C127NO3 | $1.9\times10^4$ | | Wang et al. (2017) | Q | 80, 238 |
| $C_{12}H_{19}NO_5$ | $1.7\times10^5$ | | Wang et al. (2017) | Q | 80, 239 |
| RZJFRKYKWJUXLW-UHFFFAOYSA-N | $4.2\times10^2$ | | Wang et al. (2017) | Q | 80, 240 |
| MCM:C1311NO3 | $1.7\times10^4$ | | Wang et al. (2017) | Q | 80, 238 |
| $C_{13}H_{21}NO_5$ | $1.4\times10^5$ | | Wang et al. (2017) | Q | 80, 239 |
| HIIZKUNLVYOKLJ-UHFFFAOYSA-N | $1.0\times10^4$ | | Wang et al. (2017) | Q | 80, 240 |



Table A4.6: Nitrates ($RONO_2$) (...continued)

| Substance Formula (Trivial Name) [CAS Registry Number] InChIKey | $H_s^{cp}$ (at $T^\ominus$) $\left[\dfrac{\mathrm{mol}}{\mathrm{m^3\,Pa}}\right]$ | $\dfrac{\mathrm{d}\ln H_s^{cp}}{\mathrm{d}(1/T)}$ [K] | Reference | Type | Note |
|---|---|---|---|---|---|
| MCM:C1313NO3 | $9.1\times10^6$ | | Wang et al. (2017) | Q | 80, 238 |
| $C_{13}H_{21}NO_6$ | $5.8\times10^7$ | | Wang et al. (2017) | Q | 80, 239 |
| SXMORIYKGFOKAB-UHFFFAOYSA-N | $2.2\times10^4$ | | Wang et al. (2017) | Q | 80, 240 |
| MCM:NC1313OH | $9.1\times10^6$ | | Wang et al. (2017) | Q | 80, 238 |
| $C_{13}H_{21}NO_6$ | $6.8\times10^7$ | | Wang et al. (2017) | Q | 80, 239 |
| RAUQQNONBFDDJB-UHFFFAOYSA-N | $5.5\times10^2$ | | Wang et al. (2017) | Q | 80, 240 |
| MCM:C3NO3COOOH | $5.8\times10^5$ | | Wang et al. (2017) | Q | 80, 238 |
| $C_3H_5NO_6$ | $4.0\times10^4$ | | Wang et al. (2017) | Q | 80, 239 |
| JXBSDKIVLYRFTK-UHFFFAOYSA-N | $2.0\times10^2$ | | Wang et al. (2017) | Q | 80, 240 |
| MCM:CH3COPAN | $2.6\times10^3$ | | Wang et al. (2017) | Q | 80, 238 |
| $C_3H_3NO_6$ | $2.1\times10^3$ | | Wang et al. (2017) | Q | 80, 239 |
| BNYSUSDQQIWDSZ-UHFFFAOYSA-N | $2.2\times10^{-3}$ | | Wang et al. (2017) | Q | 80, 240 |
| MCM:BUTONENO3 | $7.1$ | | Wang et al. (2017) | Q | 80, 238 |
| $C_4H_7NO_4$ | $3.5\times10^1$ | | Wang et al. (2017) | Q | 80, 239 |
| KFRUNLQRYXGRRS-UHFFFAOYSA-N | $7.4\times10^{-2}$ | | Wang et al. (2017) | Q | 80, 240 |
| MCM:CO23C4NO3 | $3.9\times10^3$ | | Wang et al. (2017) | Q | 80, 238 |
| $C_4H_5NO_5$ | $4.3\times10^3$ | | Wang et al. (2017) | Q | 80, 239 |
| RSORCZUYAJEINP-UHFFFAOYSA-N | $7.1\times10^{-2}$ | | Wang et al. (2017) | Q | 80, 240 |
| MCM:CO2C3PAN | $2.4\times10^3$ | | Wang et al. (2017) | Q | 80, 238 |
| $C_4H_5NO_6$ | $4.0\times10^3$ | | Wang et al. (2017) | Q | 80, 239 |
| VXGDXYQGFKIYIJ-UHFFFAOYSA-N | $1.2\times10^{-1}$ | | Wang et al. (2017) | Q | 80, 240 |
| MCM:CO2N3CO3H | $6.8\times10^6$ | | Wang et al. (2017) | Q | 80, 238 |
| $C_4H_5NO_7$ | $5.5\times10^5$ | | Wang et al. (2017) | Q | 80, 239 |
| PLOOSZKRSUJKKF-UHFFFAOYSA-N | $4.5\times10^1$ | | Wang et al. (2017) | Q | 80, 240 |
| MCM:CO2N3PAN | $3.1\times10^5$ | | Wang et al. (2017) | Q | 80, 238 |
| $C_4H_4N_2O_9$ | $2.3\times10^5$ | | Wang et al. (2017) | Q | 80, 239 |
| RXUYCSQEPKEYNB-UHFFFAOYSA-N | $2.3\times10^{-3}$ | | Wang et al. (2017) | Q | 80, 240 |
| 2-oxobutyl nitrate | $5.9$ | | Wang et al. (2017) | Q | 80, 238 |
| $C_4H_7NO_4$ | $5.9\times10^1$ | | Wang et al. (2017) | Q | 80, 239 |
| [138779-12-1] | $3.1\times10^{-1}$ | | Wang et al. (2017) | Q | 80, 240 |
| OVISQPFXRZOZFI-UHFFFAOYSA-N | | | | | |
| MCM:COC4NO3OOH | $5.1\times10^5$ | | Wang et al. (2017) | Q | 80, 238 |
| $C_4H_7NO_6$ | $1.8\times10^4$ | | Wang et al. (2017) | Q | 80, 239 |
| AEHFBVVGPAXMNG-UHFFFAOYSA-N | $1.0\times10^2$ | | Wang et al. (2017) | Q | 80, 240 |
| MCM:MEKANO3 | $5.9$ | | Wang et al. (2017) | Q | 80, 238 |
| $C_4H_7NO_4$ | $1.3\times10^2$ | | Wang et al. (2017) | Q | 80, 239 |
| SQFMCLNZTZWUJY-UHFFFAOYSA-N | $8.9\times10^{-1}$ | | Wang et al. (2017) | Q | 80, 240 |
| MCM:NMVK | $1.7\times10^1$ | | Wang et al. (2017) | Q | 80, 238 |
| $C_4H_5NO_4$ | $1.0\times10^2$ | | Wang et al. (2017) | Q | 80, 239 |
| YTEQPVIDLVIVHW-UHFFFAOYSA-N | $3.4\times10^{-1}$ | | Wang et al. (2017) | Q | 80, 240 |





Table A4.6: Nitrates ($RONO_2$) (. . . continued)

| Substance Formula (Trivial Name) [CAS Registry Number] InChIKey | $H_s^{cp}$ (at $T^\ominus$) $\left[\dfrac{\text{mol}}{\text{m}^3\,\text{Pa}}\right]$ | $\dfrac{\text{d}\ln H_s^{cp}}{\text{d}(1/T)}$ [K] | Reference | Type | Note |
|---|---|---|---|---|---|
| MCM:C4M2NO3ONE | 3.8 | | Wang et al. (2017) | Q | 80, 238 |
| $C_5H_9NO_4$ | 7.3 | | Wang et al. (2017) | Q | 80, 239 |
| DIMIHFOKCQZMGY-UHFFFAOYSA-N | $5.9\times10^{-2}$ | | Wang et al. (2017) | Q | 80, 240 |
| MCM:C4M3NO3ONE | 5.5 | | Wang et al. (2017) | Q | 80, 238 |
| $C_5H_9NO_4$ | $3.4\times10^1$ | | Wang et al. (2017) | Q | 80, 239 |
| SAHLZHFOPIFWKH-UHFFFAOYSA-N | $1.8\times10^{-1}$ | | Wang et al. (2017) | Q | 80, 240 |
| MCM:C4MCNO3OOH | $2.9\times10^5$ | | Wang et al. (2017) | Q | 80, 238 |
| $C_5H_9NO_6$ | $8.3\times10^4$ | | Wang et al. (2017) | Q | 80, 239 |
| AOCDDMVLVLTZNT-UHFFFAOYSA-N | $2.2\times10^1$ | | Wang et al. (2017) | Q | 80, 240 |
| MCM:C51NO324CO | $3.6\times10^3$ | | Wang et al. (2017) | Q | 80, 238 |
| $C_5H_7NO_5$ | $2.0\times10^4$ | | Wang et al. (2017) | Q | 80, 239 |
| KIDZPQUIRFWTHY-UHFFFAOYSA-N | 8.0 | | Wang et al. (2017) | Q | 80, 240 |
| MCM:C51NO32CO | 4.9 | | Wang et al. (2017) | Q | 80, 238 |
| $C_5H_9NO_4$ | $3.2\times10^1$ | | Wang et al. (2017) | Q | 80, 239 |
| TTZVKGPORDTQGG-UHFFFAOYSA-N | $2.6\times10^{-1}$ | | Wang et al. (2017) | Q | 80, 240 |
| MCM:C52NO33CO | 5.5 | | Wang et al. (2017) | Q | 80, 238 |
| $C_5H_9NO_4$ | $1.6\times10^1$ | | Wang et al. (2017) | Q | 80, 239 |
| VOCQUTRGVIAJPG-UHFFFAOYSA-N | $6.2\times10^{-2}$ | | Wang et al. (2017) | Q | 80, 240 |
| MCM:C53NO324CO | $3.7\times10^3$ | | Wang et al. (2017) | Q | 80, 238 |
| $C_5H_7NO_5$ | $7.4\times10^3$ | | Wang et al. (2017) | Q | 80, 239 |
| QYJXVKSLJJQONK-UHFFFAOYSA-N | $1.3\times10^{-1}$ | | Wang et al. (2017) | Q | 80, 240 |
| MCM:C53NO32CO | 5.5 | | Wang et al. (2017) | Q | 80, 238 |
| $C_5H_9NO_4$ | $1.7\times10^1$ | | Wang et al. (2017) | Q | 80, 239 |
| LVGYWFUEHRGRSS-UHFFFAOYSA-N | $6.9\times10^{-2}$ | | Wang et al. (2017) | Q | 80, 240 |
| MCM:C5CONO3OOH | $2.9\times10^5$ | | Wang et al. (2017) | Q | 80, 238 |
| $C_5H_9NO_6$ | $3.7\times10^3$ | | Wang et al. (2017) | Q | 80, 239 |
| ZDTBCADGPXXLKE-UHFFFAOYSA-N | $3.3\times10^1$ | | Wang et al. (2017) | Q | 80, 240 |
| MCM:C5COO2NO2 | $7.6\times10^3$ | | Wang et al. (2017) | Q | 80, 238 |
| $C_5H_5NO_6$ | $3.9\times10^4$ | | Wang et al. (2017) | Q | 80, 239 |
| BZPLAICQOBBIOR-UHFFFAOYSA-N | $4.7\times10^{-2}$ | | Wang et al. (2017) | Q | 80, 240 |
| MCM:C5NO3O4OOH | $4.8\times10^5$ | | Wang et al. (2017) | Q | 80, 238 |
| $C_5H_9NO_6$ | $4.6\times10^3$ | | Wang et al. (2017) | Q | 80, 239 |
| CASCWEIOXSCVGJ-UHFFFAOYSA-N | $1.0\times10^1$ | | Wang et al. (2017) | Q | 80, 240 |
| MCM:C5NO3OAOOH | $4.2\times10^5$ | | Wang et al. (2017) | Q | 80, 238 |
| $C_5H_9NO_6$ | $5.3\times10^5$ | | Wang et al. (2017) | Q | 80, 239 |
| FLHIUJMTRYIPLY-UHFFFAOYSA-N | $1.1\times10^3$ | | Wang et al. (2017) | Q | 80, 240 |
| MCM:C5ONO34OOH | $4.8\times10^5$ | | Wang et al. (2017) | Q | 80, 238 |
| $C_5H_9NO_6$ | $1.5\times10^5$ | | Wang et al. (2017) | Q | 80, 239 |
| LOZNRBAEPWQBFW-UHFFFAOYSA-N | $2.9\times10^1$ | | Wang et al. (2017) | Q | 80, 240 |



Table A4.6: Nitrates ($RONO_2$) (. . . continued)

| Substance Formula (Trivial Name) [CAS Registry Number] InChIKey | $H_s^{cp}$ (at $T^\ominus$) $\left[\dfrac{\text{mol}}{\text{m}^3\,\text{Pa}}\right]$ | $\dfrac{\text{d}\ln H_s^{cp}}{\text{d}(1/T)}$ [K] | Reference | Type | Note |
|---|---|---|---|---|---|
| MCM:C5PAN16 | $1.9\times10^3$ | | Wang et al. (2017) | Q | 80, 238 |
| $C_5H_7NO_6$ | $2.0\times10^3$ | | Wang et al. (2017) | Q | 80, 239 |
| IZEBTYZDWKBBFQ-UHFFFAOYSA-N | $5.9\times10^{-2}$ | | Wang et al. (2017) | Q | 80, 240 |
| MCM:C5PAN2 | $1.9\times10^3$ | | Wang et al. (2017) | Q | 80, 238 |
| $C_5H_7NO_6$ | $9.6\times10^3$ | | Wang et al. (2017) | Q | 80, 239 |
| XMORPXKVSPHXDA-UHFFFAOYSA-N | $5.1\times10^{-1}$ | | Wang et al. (2017) | Q | 80, 240 |
| MCM:C5PAN7 | $2.2\times10^3$ | | Wang et al. (2017) | Q | 80, 238 |
| $C_5H_7NO_6$ | $1.5\times10^3$ | | Wang et al. (2017) | Q | 80, 239 |
| VONRVKPFQAOOPX-UHFFFAOYSA-N | $3.0\times10^{-2}$ | | Wang et al. (2017) | Q | 80, 240 |
| MCM:C5PAN9 | $1.3\times10^6$ | | Wang et al. (2017) | Q | 80, 238 |
| $C_5H_5NO_7$ | $2.5\times10^5$ | | Wang et al. (2017) | Q | 80, 239 |
| HJZPHZLLKGPYCO-UHFFFAOYSA-N | $5.3\times10^{-2}$ | | Wang et al. (2017) | Q | 80, 240 |
| MCM:DIEKBNO3 | $4.9$ | | Wang et al. (2017) | Q | 80, 238 |
| $C_5H_9NO_4$ | $5.5\times10^1$ | | Wang et al. (2017) | Q | 80, 239 |
| WKTFKAAKLZJYGP-UHFFFAOYSA-N | $5.9\times10^{-1}$ | | Wang et al. (2017) | Q | 80, 240 |
| MCM:INANCOCO3H | $4.4\times10^8$ | | Wang et al. (2017) | Q | 80, 238 |
| $C_5H_6N_2O_{10}$ | $1.4\times10^7$ | | Wang et al. (2017) | Q | 80, 239 |
| OKQCOTADQFFQES-UHFFFAOYSA-N | $1.2$ | | Wang et al. (2017) | Q | 80, 240 |
| MCM:INANCOPAN | $1.9\times10^7$ | | Wang et al. (2017) | Q | 80, 238 |
| $C_5H_5N_3O_{12}$ | $5.3\times10^6$ | | Wang et al. (2017) | Q | 80, 239 |
| OHHXWPYTKFCNFA-UHFFFAOYSA-N | $8.0\times10^{-5}$ | | Wang et al. (2017) | Q | 80, 240 |
| MCM:MIPKBNO3 | $5.5$ | | Wang et al. (2017) | Q | 80, 238 |
| $C_5H_9NO_4$ | $5.5\times10^1$ | | Wang et al. (2017) | Q | 80, 239 |
| PNKPSVUHDDVFRC-UHFFFAOYSA-N | $6.2\times10^{-1}$ | | Wang et al. (2017) | Q | 80, 240 |
| MCM:MPRKNO3 | $5.5$ | | Wang et al. (2017) | Q | 80, 238 |
| $C_5H_9NO_4$ | $3.9\times10^1$ | | Wang et al. (2017) | Q | 80, 239 |
| GMHWHQCXUOCSQQ-UHFFFAOYSA-N | $4.6\times10^{-1}$ | | Wang et al. (2017) | Q | 80, 240 |
| MCM:C3COCPAN | $1.6\times10^3$ | | Wang et al. (2017) | Q | 80, 238 |
| $C_6H_9NO_6$ | $1.1\times10^3$ | | Wang et al. (2017) | Q | 80, 239 |
| QAGGXZNGFPYUOO-UHFFFAOYSA-N | $4.9\times10^{-2}$ | | Wang et al. (2017) | Q | 80, 240 |
| MCM:C4MCODBPAN | $5.1\times10^3$ | | Wang et al. (2017) | Q | 80, 238 |
| $C_6H_7NO_6$ | $3.3\times10^4$ | | Wang et al. (2017) | Q | 80, 239 |
| OBEVNDXPXGMJPT-UHFFFAOYSA-N | $1.4\times10^{-2}$ | | Wang et al. (2017) | Q | 80, 240 |
| MCM:C62NO335CO | $3.4\times10^3$ | | Wang et al. (2017) | Q | 80, 238 |
| $C_6H_9NO_5$ | $6.0\times10^3$ | | Wang et al. (2017) | Q | 80, 239 |
| ABDNFJAHWTWRFH-UHFFFAOYSA-N | $3.9$ | | Wang et al. (2017) | Q | 80, 240 |
| MCM:C62NO33CO | $4.6$ | | Wang et al. (2017) | Q | 80, 238 |
| $C_6H_{11}NO_4$ | $9.8$ | | Wang et al. (2017) | Q | 80, 239 |
| OBVLXHVNJQTRAU-UHFFFAOYSA-N | $4.5\times10^{-2}$ | | Wang et al. (2017) | Q | 80, 240 |



Table A4.6: Nitrates ($RONO_2$) (...continued)

| Substance<br>Formula<br>(Trivial Name)<br>[CAS Registry Number]<br>InChIKey | $H_s^{cp}$<br>(at $T^{\ominus}$)<br>$\left[\dfrac{\text{mol}}{\text{m}^3\,\text{Pa}}\right]$ | $\dfrac{\mathrm{d}\ln H_s^{cp}}{\mathrm{d}(1/T)}$<br><br>[K] | Reference | Type | Note |
|---|---|---|---|---|---|
| MCM:C63NO32CO | 4.6 | | Wang et al. (2017) | Q | 80, 238 |
| $C_6H_{11}NO_4$ | $1.1\times10^1$ | | Wang et al. (2017) | Q | 80, 239 |
| QSKMGJAWRGUYGL-UHFFFAOYSA-N | $5.5\times10^{-2}$ | | Wang et al. (2017) | Q | 80, 240 |
| MCM:C64NO335CO | $3.4\times10^3$ | | Wang et al. (2017) | Q | 80, 238 |
| $C_6H_9NO_5$ | $4.2\times10^3$ | | Wang et al. (2017) | Q | 80, 239 |
| CHGSCHAFWLLPDM-UHFFFAOYSA-N | $8.0\times10^{-2}$ | | Wang et al. (2017) | Q | 80, 240 |
| MCM:C66NO35CO | 4.3 | | Wang et al. (2017) | Q | 80, 238 |
| $C_6H_{11}NO_4$ | $2.2\times10^1$ | | Wang et al. (2017) | Q | 80, 239 |
| NNYDFJRPKYRBEY-UHFFFAOYSA-N | $1.4\times10^{-1}$ | | Wang et al. (2017) | Q | 80, 240 |
| MCM:C6CO134PAN | $1.1\times10^6$ | | Wang et al. (2017) | Q | 80, 238 |
| $C_6H_7NO_7$ | $1.4\times10^5$ | | Wang et al. (2017) | Q | 80, 239 |
| OIZONYBQGYIWDV-UHFFFAOYSA-N | $2.2\times10^{-2}$ | | Wang et al. (2017) | Q | 80, 240 |
| MCM:C6CONO3OOH | $3.9\times10^5$ | | Wang et al. (2017) | Q | 80, 238 |
| $C_6H_{11}NO_6$ | $1.3\times10^5$ | | Wang et al. (2017) | Q | 80, 239 |
| JAMSWIXAFBYGGR-UHFFFAOYSA-N | $3.8\times10^2$ | | Wang et al. (2017) | Q | 80, 240 |
| MCM:C6DCARBPAN | $7.1\times10^3$ | | Wang et al. (2017) | Q | 80, 238 |
| $C_6H_7NO_6$ | $1.8\times10^4$ | | Wang et al. (2017) | Q | 80, 239 |
| QTAROYORLXCHIO-UHFFFAOYSA-N | $3.0\times10^{-2}$ | | Wang et al. (2017) | Q | 80, 240 |
| MCM:C6NO324CO | $2.9\times10^3$ | | Wang et al. (2017) | Q | 80, 238 |
| $C_6H_9NO_5$ | $1.1\times10^4$ | | Wang et al. (2017) | Q | 80, 239 |
| ZLHGCQIPJTVAEP-UHFFFAOYSA-N | 4.4 | | Wang et al. (2017) | Q | 80, 240 |
| MCM:C6NO3COOOH | $3.3\times10^5$ | | Wang et al. (2017) | Q | 80, 238 |
| $C_6H_{11}NO_6$ | $2.6\times10^5$ | | Wang et al. (2017) | Q | 80, 239 |
| HQTZSBFILMVSJZ-UHFFFAOYSA-N | $5.9\times10^2$ | | Wang et al. (2017) | Q | 80, 240 |
| MCM:C6PAN12 | $1.7\times10^3$ | | Wang et al. (2017) | Q | 80, 238 |
| $C_6H_9NO_6$ | $3.4\times10^3$ | | Wang et al. (2017) | Q | 80, 239 |
| LSIJQLAUAXHOJI-UHFFFAOYSA-N | $2.0\times10^{-1}$ | | Wang et al. (2017) | Q | 80, 240 |
| MCM:C6PAN16 | $1.2\times10^3$ | | Wang et al. (2017) | Q | 80, 238 |
| $C_6H_9NO_6$ | $5.3\times10^2$ | | Wang et al. (2017) | Q | 80, 239 |
| JDUNWYCOIXFXGK-UHFFFAOYSA-N | $1.8\times10^{-2}$ | | Wang et al. (2017) | Q | 80, 240 |
| MCM:C6PAN2 | $1.6\times10^3$ | | Wang et al. (2017) | Q | 80, 238 |
| $C_6H_9NO_6$ | $4.9\times10^3$ | | Wang et al. (2017) | Q | 80, 239 |
| BSRDZZSTWMTZHH-UHFFFAOYSA-N | $4.0\times10^{-1}$ | | Wang et al. (2017) | Q | 80, 240 |
| MCM:C6PAN5 | $1.7\times10^3$ | | Wang et al. (2017) | Q | 80, 238 |
| $C_6H_9NO_6$ | $3.4\times10^3$ | | Wang et al. (2017) | Q | 80, 239 |
| AYORGEAGFIRFHW-UHFFFAOYSA-N | $4.1\times10^{-1}$ | | Wang et al. (2017) | Q | 80, 240 |
| MCM:C6PAN6 | $1.7\times10^3$ | | Wang et al. (2017) | Q | 80, 238 |
| $C_6H_9NO_6$ | $6.6\times10^2$ | | Wang et al. (2017) | Q | 80, 239 |
| STXATUQYWORJHN-UHFFFAOYSA-N | $1.5\times10^{-2}$ | | Wang et al. (2017) | Q | 80, 240 |





Table A4.6: Nitrates ($RONO_2$) (...continued)

| Substance Formula (Trivial Name) [CAS Registry Number] InChIKey | $H_s^{cp}$ (at $T^{\ominus}$) $\left[\dfrac{\mathrm{mol}}{\mathrm{m^3\,Pa}}\right]$ | $\dfrac{\mathrm{d}\ln H_s^{cp}}{\mathrm{d}(1/T)}$ [K] | Reference | Type | Note |
|---|---|---|---|---|---|
| MCM:C6PAN7 $C_6H_7NO_7$ FEUNFERSGUKVTQ-UHFFFAOYSA-N | $1.3\times10^6$ $1.0\times10^5$ $2.7\times10^{-2}$ | | Wang et al. (2017) Wang et al. (2017) Wang et al. (2017) | Q Q Q | 80, 238 80, 239 80, 240 |
| MCM:CONO3C6OOH $C_6H_{11}NO_6$ BPHZXGKZHPQLPB-UHFFFAOYSA-N | $3.9\times10^5$ $7.3\times10^4$ $1.5\times10^1$ | | Wang et al. (2017) Wang et al. (2017) Wang et al. (2017) | Q Q Q | 80, 238 80, 239 80, 240 |
| MCM:CYHXONANO3 $C_6H_9NO_4$ VLCBNIODXGFPBM-UHFFFAOYSA-N | $1.4\times10^1$ $3.6\times10^2$ $9.8$ | | Wang et al. (2017) Wang et al. (2017) Wang et al. (2017) | Q Q Q | 80, 238 80, 239 80, 240 |
| MCM:ECO3PAN $C_6H_5NO_8$ GLZSBLKUFXIYNN-UHFFFAOYSA-N | $7.8\times10^8$ $4.0\times10^7$ $3.4\times10^{-4}$ | | Wang et al. (2017) Wang et al. (2017) Wang et al. (2017) | Q Q Q | 80, 238 80, 239 80, 240 |
| MCM:EIPKBNO3 $C_6H_{11}NO_4$ QPLZWEVCYZLYTL-UHFFFAOYSA-N | $4.6$ $2.7\times10^1$ $2.7\times10^{-1}$ | | Wang et al. (2017) Wang et al. (2017) Wang et al. (2017) | Q Q Q | 80, 238 80, 239 80, 240 |
| MCM:HEX2ONANO3 $C_6H_{11}NO_4$ OZSNHPHDGAJMHG-UHFFFAOYSA-N | $4.6$ $2.2\times10^1$ $2.7\times10^{-1}$ | | Wang et al. (2017) Wang et al. (2017) Wang et al. (2017) | Q Q Q | 80, 238 80, 239 80, 240 |
| MCM:HEX2ONBNO3 $C_6H_{11}NO_4$ JYVFNKYOALTNTA-UHFFFAOYSA-N | $4.6$ $3.0\times10^1$ $1.0$ | | Wang et al. (2017) Wang et al. (2017) Wang et al. (2017) | Q Q Q | 80, 238 80, 239 80, 240 |
| MCM:HEX3ONANO3 $C_6H_{11}NO_4$ KTYGBCIWLHIEKP-UHFFFAOYSA-N | $4.6$ $2.0\times10^1$ $4.0\times10^{-1}$ | | Wang et al. (2017) Wang et al. (2017) Wang et al. (2017) | Q Q Q | 80, 238 80, 239 80, 240 |
| MCM:HEX3ONDNO3 $C_6H_{11}NO_4$ PRTKDFINSKNZSU-UHFFFAOYSA-N | $4.3$ $3.6\times10^1$ $6.3\times10^{-1}$ | | Wang et al. (2017) Wang et al. (2017) Wang et al. (2017) | Q Q Q | 80, 238 80, 239 80, 240 |
| MCM:M2BKANO3 $C_6H_{11}NO_4$ NYAAKBDXIQFYDD-UHFFFAOYSA-N | $5.1$ $2.6\times10^1$ $3.6\times10^{-1}$ | | Wang et al. (2017) Wang et al. (2017) Wang et al. (2017) | Q Q Q | 80, 238 80, 239 80, 240 |
| MCM:MC4CODBPAN $C_6H_7NO_6$ WHFHKQLMYAOVSY-UHFFFAOYSA-N | $5.1\times10^3$ $3.3\times10^4$ $2.2\times10^{-2}$ | | Wang et al. (2017) Wang et al. (2017) Wang et al. (2017) | Q Q Q | 80, 238 80, 239 80, 240 |
| MCM:MIBKANO3 $C_6H_{11}NO_4$ PKJJSCYTBLHEMI-UHFFFAOYSA-N | $3.0$ $1.5\times10^1$ $1.7\times10^{-1}$ | | Wang et al. (2017) Wang et al. (2017) Wang et al. (2017) | Q Q Q | 80, 238 80, 239 80, 240 |
| MCM:MTBKNO3 $C_6H_{11}NO_4$ OYSBYTCKEAIMOW-UHFFFAOYSA-N | $3.0$ $2.6\times10^1$ $5.4\times10^{-1}$ | | Wang et al. (2017) Wang et al. (2017) Wang et al. (2017) | Q Q Q | 80, 238 80, 239 80, 240 |





Table A4.6: Nitrates ($RONO_2$) (...continued)

| Substance<br>Formula<br>(Trivial Name)<br>[CAS Registry Number]<br>InChIKey | $H_s^{cp}$<br>(at $T^\ominus$)<br>$\left[\dfrac{mol}{m^3\,Pa}\right]$ | $\dfrac{d\ln H_s^{cp}}{d(1/T)}$<br><br>[K] | Reference | Type | Note |
|---|---|---|---|---|---|
| MCM:NBZQOOH<br>$C_6H_5NO_7$<br>NXGPVQDYDGLWQQ-UHFFFAOYSA-N | $3.2\times10^9$<br>$5.6\times10^8$<br>$1.7\times10^3$ | | Wang et al. (2017)<br>Wang et al. (2017)<br>Wang et al. (2017) | Q<br>Q<br>Q | 80, 238<br>80, 239<br>80, 240 |
| MCM:C627PAN<br>$C_7H_9NO_7$<br>FQVTWXYWQQSSRX-UHFFFAOYSA-N | $9.1\times10^5$<br>$3.0\times10^6$<br>$2.1\times10^1$ | | Wang et al. (2017)<br>Wang et al. (2017)<br>Wang et al. (2017) | Q<br>Q<br>Q | 80, 238<br>80, 239<br>80, 240 |
| MCM:C7ADCPAN<br>$C_7H_9NO_6$<br>OFKNKTFVVCSQBW-UHFFFAOYSA-N | $3.5\times10^3$<br>$4.3\times10^4$<br>$8.1\times10^{-2}$ | | Wang et al. (2017)<br>Wang et al. (2017)<br>Wang et al. (2017) | Q<br>Q<br>Q | 80, 238<br>80, 239<br>80, 240 |
| MCM:C7DCPAN<br>$C_7H_9NO_6$<br>ZGBUEZLBCHGZPP-UHFFFAOYSA-N | $5.5\times10^3$<br>$9.6\times10^3$<br>$1.4\times10^{-2}$ | | Wang et al. (2017)<br>Wang et al. (2017)<br>Wang et al. (2017) | Q<br>Q<br>Q | 80, 238<br>80, 239<br>80, 240 |
| MCM:C7DDCPAN<br>$C_7H_9NO_6$<br>WCHGYJHJZMHIAR-UHFFFAOYSA-N | $4.6\times10^3$<br>$1.4\times10^4$<br>$5.5\times10^{-3}$ | | Wang et al. (2017)<br>Wang et al. (2017)<br>Wang et al. (2017) | Q<br>Q<br>Q | 80, 238<br>80, 239<br>80, 240 |
| MCM:C7PAN3<br>$C_7H_7NO_8$<br>XMNXEIJPBOBVEP-UHFFFAOYSA-N | $6.3\times10^8$<br>$1.7\times10^7$<br>$1.4$ | | Wang et al. (2017)<br>Wang et al. (2017)<br>Wang et al. (2017) | Q<br>Q<br>Q | 80, 238<br>80, 239<br>80, 240 |
| MCM:IC7DCPAN<br>$C_7H_9NO_6$<br>NZLPHNCQTJFNGH-UHFFFAOYSA-N | $6.3\times10^3$<br>$1.0\times10^4$<br>$8.1\times10^{-3}$ | | Wang et al. (2017)<br>Wang et al. (2017)<br>Wang et al. (2017) | Q<br>Q<br>Q | 80, 238<br>80, 239<br>80, 240 |
| MCM:NC71CO<br>$C_7H_7NO_6$<br>CDHAVACRDDMHGZ-UHFFFAOYSA-N | $3.9\times10^6$<br>$2.5\times10^7$<br>$1.2\times10^3$ | | Wang et al. (2017)<br>Wang et al. (2017)<br>Wang et al. (2017) | Q<br>Q<br>Q | 80, 238<br>80, 239<br>80, 240 |
| MCM:NC71OOH<br>$C_7H_9NO_7$<br>CIJBWMVCSXGACJ-UHFFFAOYSA-N | $4.5\times10^8$<br>$1.6\times10^9$<br>$5.6\times10^5$ | | Wang et al. (2017)<br>Wang et al. (2017)<br>Wang et al. (2017) | Q<br>Q<br>Q | 80, 238<br>80, 239<br>80, 240 |
| MCM:NC72OOH<br>$C_7H_7NO_8$<br>RYHDNZNGTUJGAD-UHFFFAOYSA-N | $3.0\times10^{11}$<br>$3.6\times10^{10}$<br>$5.8\times10^5$ | | Wang et al. (2017)<br>Wang et al. (2017)<br>Wang et al. (2017) | Q<br>Q<br>Q | 80, 238<br>80, 239<br>80, 240 |
| MCM:NPTLQOOH<br>$C_7H_7NO_7$<br>WIZJVJIMHIROQF-UHFFFAOYSA-N | $2.2\times10^9$<br>$4.8\times10^8$<br>$6.0\times10^2$ | | Wang et al. (2017)<br>Wang et al. (2017)<br>Wang et al. (2017) | Q<br>Q<br>Q | 80, 238<br>80, 239<br>80, 240 |
| MCM:C5DBEPAN<br>$C_8H_9NO_7$<br>OYOZKJOWUFGKRQ-UHFFFAOYSA-N | $2.5\times10^6$<br>$1.1\times10^6$<br>$9.6\times10^{-3}$ | | Wang et al. (2017)<br>Wang et al. (2017)<br>Wang et al. (2017) | Q<br>Q<br>Q | 80, 238<br>80, 239<br>80, 240 |
| MCM:C5EDBPAN<br>$C_8H_9NO_7$<br>KNPMBKOCKIUXFV-UHFFFAOYSA-N | $2.5\times10^6$<br>$1.1\times10^6$<br>$7.6\times10^{-3}$ | | Wang et al. (2017)<br>Wang et al. (2017)<br>Wang et al. (2017) | Q<br>Q<br>Q | 80, 238<br>80, 239<br>80, 240 |





Table A4.6: Nitrates ($RONO_2$) (. . . continued)

| Substance Formula (Trivial Name) [CAS Registry Number] InChIKey | $H_s^{cp}$ (at $T^\ominus$) $\left[\dfrac{\text{mol}}{\text{m}^3\,\text{Pa}}\right]$ | $\dfrac{\mathrm{d}\ln H_s^{cp}}{\mathrm{d}(1/T)}$ [K] | Reference | Type | Note |
|---|---|---|---|---|---|
| MCM:C727PAN | $8.5\times10^5$ | | Wang et al. (2017) | Q | 80, 238 |
| $C_8H_{11}NO_7$ | $6.9\times10^5$ | | Wang et al. (2017) | Q | 80, 239 |
| WBGZRMCIPBEEGH-UHFFFAOYSA-N | 4.1 | | Wang et al. (2017) | Q | 80, 240 |
| MCM:C7CODBPAN | $3.7\times10^3$ | | Wang et al. (2017) | Q | 80, 238 |
| $C_8H_{11}NO_6$ | $7.6\times10^3$ | | Wang et al. (2017) | Q | 80, 239 |
| QVHZZCVTEAGIPK-UHFFFAOYSA-N | $3.8\times10^{-3}$ | | Wang et al. (2017) | Q | 80, 240 |
| MCM:C817NO3 | $3.9\times10^4$ | 12000 | Wieser et al. (2023) | Q | 437 |
| $C_8H_{13}NO_5$ | $2.2\times10^3$ | | Wang et al. (2017) | Q | 80, 238 |
| ZONZWPKPCIWKRN-UHFFFAOYSA-N | $3.8\times10^4$ | | Wang et al. (2017) | Q | 80, 239 |
| | $3.1\times10^2$ | | Wang et al. (2017) | Q | 80, 240 |
| MCM:NMXYQOOH | $1.2\times10^9$ | | Wang et al. (2017) | Q | 80, 238 |
| $C_8H_9NO_7$ | $9.6\times10^7$ | | Wang et al. (2017) | Q | 80, 239 |
| MDRQMTCFLWPCNU-UHFFFAOYSA-N | $4.0\times10^2$ | | Wang et al. (2017) | Q | 80, 240 |
| MCM:NOXYQOOH | $1.5\times10^9$ | | Wang et al. (2017) | Q | 80, 238 |
| $C_8H_9NO_7$ | $3.8\times10^8$ | | Wang et al. (2017) | Q | 80, 239 |
| SFYCEAPRSKIFCS-UHFFFAOYSA-N | $4.2\times10^2$ | | Wang et al. (2017) | Q | 80, 240 |
| MCM:NPEBQOOH | $2.0\times10^9$ | | Wang et al. (2017) | Q | 80, 238 |
| $C_8H_9NO_7$ | $2.6\times10^8$ | | Wang et al. (2017) | Q | 80, 239 |
| HZFUWULOZSGVHH-UHFFFAOYSA-N | $5.5\times10^2$ | | Wang et al. (2017) | Q | 80, 240 |
| MCM:NPXYQOOH | $1.2\times10^9$ | | Wang et al. (2017) | Q | 80, 238 |
| $C_8H_9NO_7$ | $9.3\times10^7$ | | Wang et al. (2017) | Q | 80, 239 |
| LMPSHWMXIGVDOI-UHFFFAOYSA-N | $2.3\times10^2$ | | Wang et al. (2017) | Q | 80, 240 |
| MCM:C816PAN | $2.3\times10^3$ | | Wang et al. (2017) | Q | 80, 238 |
| $C_9H_{13}NO_6$ | $1.4\times10^3$ | | Wang et al. (2017) | Q | 80, 239 |
| JXXOSNUALVJMQL-UHFFFAOYSA-N | $2.0\times10^{-1}$ | | Wang et al. (2017) | Q | 80, 240 |
| MCM:C817PAN | $6.9\times10^5$ | | Wang et al. (2017) | Q | 80, 238 |
| $C_9H_{13}NO_7$ | $1.6\times10^6$ | | Wang et al. (2017) | Q | 80, 239 |
| AGDIJRUETYYZKI-UHFFFAOYSA-N | $3.2\times10^2$ | | Wang et al. (2017) | Q | 80, 240 |
| MCM:C827PAN | $4.8\times10^5$ | | Wang et al. (2017) | Q | 80, 238 |
| $C_9H_{13}NO_7$ | $5.3\times10^5$ | | Wang et al. (2017) | Q | 80, 239 |
| QMDLSNSKOZAILD-UHFFFAOYSA-N | 2.3 | | Wang et al. (2017) | Q | 80, 240 |
| MCM:C828PAN | $3.6\times10^8$ | | Wang et al. (2017) | Q | 80, 238 |
| $C_9H_{11}NO_8$ | $1.1\times10^7$ | | Wang et al. (2017) | Q | 80, 239 |
| LCFHBKSAGMTDCX-UHFFFAOYSA-N | 1.1 | | Wang et al. (2017) | Q | 80, 240 |
| MCM:C88PAN | $1.7\times10^6$ | | Wang et al. (2017) | Q | 80, 238 |
| $C_9H_{11}NO_7$ | $1.4\times10^7$ | | Wang et al. (2017) | Q | 80, 239 |
| KLGKHKJTYFCRCN-UHFFFAOYSA-N | $3.6\times10^1$ | | Wang et al. (2017) | Q | 80, 240 |
| MCM:C917NO3 | $4.3\times10^3$ | | Wang et al. (2017) | Q | 80, 238 |
| $C_9H_{13}NO_5$ | $2.5\times10^5$ | | Wang et al. (2017) | Q | 80, 239 |
| KVWUPUMJPKRVQS-UHFFFAOYSA-N | $2.3\times10^3$ | | Wang et al. (2017) | Q | 80, 240 |



Table A4.6: Nitrates ($RONO_2$) (...continued)

| Substance Formula (Trivial Name) [CAS Registry Number] InChIKey | $H_s^{cp}$ (at $T^\ominus$) $\left[\dfrac{\mathrm{mol}}{\mathrm{m^3\,Pa}}\right]$ | $\dfrac{\mathrm{d}\ln H_s^{cp}}{\mathrm{d}(1/T)}$ [K] | Reference | Type | Note |
|---|---|---|---|---|---|
| MCM:C923NO3 | $3.2\times10^1$ | 11000 | Wieser et al. (2023) | Q | 437 |
| $C_9H_{15}NO_4$ | 6.0 | | Wang et al. (2017) | Q | 80, 238 |
| JEWJQSHMOKPQHI-UHFFFAOYSA-N | $2.5\times10^1$ | | Wang et al. (2017) | Q | 80, 239 |
| | 4.1 | | Wang et al. (2017) | Q | 80, 240 |
| MCM:C928NO3 | $1.2\times10^3$ | | Wang et al. (2017) | Q | 80, 238 |
| $C_9H_{15}NO_5$ | $3.2\times10^4$ | | Wang et al. (2017) | Q | 80, 239 |
| HCHMIBGRYQMLLZ-UHFFFAOYSA-N | $6.0\times10^1$ | | Wang et al. (2017) | Q | 80, 240 |
| MCM:C96NO3 | 6.6 | | Wang et al. (2017) | Q | 80, 238 |
| $C_9H_{15}NO_4$ | $5.0\times10^1$ | | Wang et al. (2017) | Q | 80, 239 |
| MMTOEOITRCIZDG-UHFFFAOYSA-N | 7.8 | | Wang et al. (2017) | Q | 80, 240 |
| MCM:C9DCNO3 | $1.6\times10^4$ | | Wang et al. (2017) | Q | 80, 238 |
| $C_9H_{11}NO_5$ | $7.3\times10^5$ | | Wang et al. (2017) | Q | 80, 239 |
| TURIVFFHDFEFMU-UHFFFAOYSA-N | $3.2\times10^2$ | | Wang et al. (2017) | Q | 80, 240 |
| MCM:C9PAN2 | $2.6\times10^3$ | | Wang et al. (2017) | Q | 80, 238 |
| $C_9H_{13}NO_6$ | $3.2\times10^3$ | | Wang et al. (2017) | Q | 80, 239 |
| VXDDXZNSSANCGN-UHFFFAOYSA-N | $8.7\times10^{-1}$ | | Wang et al. (2017) | Q | 80, 240 |
| MCM:NIPRBQOOH | $1.8\times10^9$ | | Wang et al. (2017) | Q | 80, 238 |
| $C_9H_{11}NO_7$ | $1.8\times10^8$ | | Wang et al. (2017) | Q | 80, 239 |
| KGZDDJNUZZWZIJ-UHFFFAOYSA-N | $2.5\times10^2$ | | Wang et al. (2017) | Q | 80, 240 |
| MCM:NLMKAOOH | $5.0\times10^5$ | | Wang et al. (2017) | Q | 80, 238 |
| $C_9H_{15}NO_6$ | $8.9\times10^5$ | | Wang et al. (2017) | Q | 80, 239 |
| ZOTFHCKOVMRARL-UHFFFAOYSA-N | $5.3\times10^3$ | | Wang et al. (2017) | Q | 80, 240 |
| MCM:NMETLQOOH | $1.1\times10^9$ | | Wang et al. (2017) | Q | 80, 238 |
| $C_9H_{11}NO_7$ | $5.0\times10^7$ | | Wang et al. (2017) | Q | 80, 239 |
| FSVFAZAPMWXECG-UHFFFAOYSA-N | $1.4\times10^2$ | | Wang et al. (2017) | Q | 80, 240 |
| MCM:NOETLQOOH | $1.2\times10^9$ | | Wang et al. (2017) | Q | 80, 238 |
| $C_9H_{11}NO_7$ | $2.1\times10^8$ | | Wang et al. (2017) | Q | 80, 239 |
| GHGZYHLEQYQZHR-UHFFFAOYSA-N | $3.8\times10^2$ | | Wang et al. (2017) | Q | 80, 240 |
| MCM:NOPINANO3 | $2.3\times10^1$ | | Wang et al. (2017) | Q | 80, 238 |
| $C_9H_{13}NO_4$ | $2.3\times10^2$ | | Wang et al. (2017) | Q | 80, 239 |
| JEOFIBHUYPQAJL-UHFFFAOYSA-N | $1.4\times10^1$ | | Wang et al. (2017) | Q | 80, 240 |
| MCM:NOPINBNO3 | $2.3\times10^1$ | | Wang et al. (2017) | Q | 80, 238 |
| $C_9H_{13}NO_4$ | $3.1\times10^2$ | | Wang et al. (2017) | Q | 80, 239 |
| QGVNTXJLRAHBMW-UHFFFAOYSA-N | $1.4\times10^1$ | | Wang et al. (2017) | Q | 80, 240 |
| MCM:NOPINCNO3 | $1.3\times10^1$ | | Wang et al. (2017) | Q | 80, 238 |
| $C_9H_{13}NO_4$ | $8.1\times10^1$ | | Wang et al. (2017) | Q | 80, 239 |
| CLMUTFVSCXCGKI-UHFFFAOYSA-N | $1.9\times10^1$ | | Wang et al. (2017) | Q | 80, 240 |
| MCM:NPETLQOOH | $1.1\times10^9$ | | Wang et al. (2017) | Q | 80, 238 |
| $C_9H_{11}NO_7$ | $4.8\times10^7$ | | Wang et al. (2017) | Q | 80, 239 |
| IMJMJNZTMWKOFO-UHFFFAOYSA-N | $2.6\times10^2$ | | Wang et al. (2017) | Q | 80, 240 |





Table A4.6: Nitrates ($RONO_2$) (...continued)

| Substance Formula (Trivial Name) [CAS Registry Number] InChIKey | $H_s^{cp}$ (at $T^\ominus$) $\left[\dfrac{mol}{m^3\,Pa}\right]$ | $\dfrac{d\ln H_s^{cp}}{d(1/T)}$ [K] | Reference | Type | Note |
|---|---|---|---|---|---|
| MCM:NPPRBQOOH | $1.6\times10^9$ | | Wang et al. (2017) | Q | 80, 238 |
| $C_9H_{11}NO_7$ | $1.8\times10^8$ | | Wang et al. (2017) | Q | 80, 239 |
| MMAZKJNYHBQGDL-UHFFFAOYSA-N | $1.7\times10^2$ | | Wang et al. (2017) | Q | 80, 240 |
| MCM:NTM124QOOH | $8.1\times10^8$ | | Wang et al. (2017) | Q | 80, 238 |
| $C_9H_{11}NO_7$ | $6.9\times10^7$ | | Wang et al. (2017) | Q | 80, 239 |
| ZMXXAPJGXNLYQU-UHFFFAOYSA-N | $5.3\times10^1$ | | Wang et al. (2017) | Q | 80, 240 |
| MCM:C1011NO3 | 5.4 | | Wang et al. (2017) | Q | 80, 238 |
| $C_{10}H_{17}NO_4$ | $2.5\times10^1$ | | Wang et al. (2017) | Q | 80, 239 |
| KLQCUGBDZZARMU-UHFFFAOYSA-N | $1.0\times10^1$ | | Wang et al. (2017) | Q | 80, 240 |
| MCM:C10PAN2 | $2.1\times10^3$ | | Wang et al. (2017) | Q | 80, 238 |
| $C_{10}H_{15}NO_6$ | $2.6\times10^3$ | | Wang et al. (2017) | Q | 80, 239 |
| XGWGKMXMXMPDAK-UHFFFAOYSA-N | 4.7 | | Wang et al. (2017) | Q | 80, 240 |
| MCM:C923PAN | $2.0\times10^3$ | | Wang et al. (2017) | Q | 80, 238 |
| $C_{10}H_{15}NO_6$ | $1.0\times10^3$ | | Wang et al. (2017) | Q | 80, 239 |
| ZVYIPTRRRFMSSU-UHFFFAOYSA-N | 2.3 | | Wang et al. (2017) | Q | 80, 240 |
| MCM:C928PAN | $3.8\times10^5$ | | Wang et al. (2017) | Q | 80, 238 |
| $C_{10}H_{15}NO_7$ | $1.5\times10^6$ | | Wang et al. (2017) | Q | 80, 239 |
| MIISGCVMOACDOR-UHFFFAOYSA-N | $2.0\times10^1$ | | Wang et al. (2017) | Q | 80, 240 |
| MCM:NC101CO | $1.4\times10^1$ | | Wang et al. (2017) | Q | 80, 238 |
| $C_{10}H_{15}NO_4$ | $4.4\times10^1$ | | Wang et al. (2017) | Q | 80, 239 |
| BCIULZBFFUODJR-UHFFFAOYSA-N | $4.2\times10^{-1}$ | | Wang et al. (2017) | Q | 80, 240 |
| MCM:NC101OOH | $6.0\times10^5$ | | Wang et al. (2017) | Q | 80, 238 |
| $C_{10}H_{15}NO_6$ | $3.4\times10^5$ | | Wang et al. (2017) | Q | 80, 239 |
| AQVSRLWLNUXIOL-UHFFFAOYSA-N | $5.8\times10^3$ | | Wang et al. (2017) | Q | 80, 240 |
| MCM:NC102OOH | $1.9\times10^8$ | | Wang et al. (2017) | Q | 80, 238 |
| $C_{10}H_{15}NO_7$ | $4.0\times10^8$ | | Wang et al. (2017) | Q | 80, 239 |
| ZNOCONLMTXMLFU-UHFFFAOYSA-N | $4.1\times10^4$ | | Wang et al. (2017) | Q | 80, 240 |
| MCM:C1011PAN | $1.9\times10^3$ | | Wang et al. (2017) | Q | 80, 238 |
| $C_{11}H_{17}NO_6$ | $2.2\times10^3$ | | Wang et al. (2017) | Q | 80, 239 |
| PRDFPHJPDVZXHH-UHFFFAOYSA-N | 2.9 | | Wang et al. (2017) | Q | 80, 240 |
| MCM:C131NO3 | $2.3\times10^3$ | | Wang et al. (2017) | Q | 80, 238 |
| $C_{13}H_{21}NO_5$ | $3.5\times10^4$ | | Wang et al. (2017) | Q | 80, 239 |
| HFRDBZMTZNPZMV-UHFFFAOYSA-N | $2.2\times10^3$ | | Wang et al. (2017) | Q | 80, 240 |
| MCM:C131PAN | $7.3\times10^5$ | | Wang et al. (2017) | Q | 80, 238 |
| $C_{14}H_{21}NO_7$ | $2.0\times10^6$ | | Wang et al. (2017) | Q | 80, 239 |
| YHMAABVAIQVHRP-UHFFFAOYSA-N | $1.3\times10^3$ | | Wang et al. (2017) | Q | 80, 240 |
| MCM:C141NO3 | 5.9 | | Wang et al. (2017) | Q | 80, 238 |
| $C_{14}H_{23}NO_4$ | $3.3\times10^1$ | | Wang et al. (2017) | Q | 80, 239 |
| JPXQKVXALDMDKF-UHFFFAOYSA-N | $3.6\times10^1$ | | Wang et al. (2017) | Q | 80, 240 |



Table A4.6: Nitrates ($RONO_2$) (...continued)

| Substance<br>Formula<br>(Trivial Name)<br>[CAS Registry Number]<br>InChIKey | $H_s^{cp}$<br>(at $T^\ominus$)<br>$\left[\dfrac{\text{mol}}{\text{m}^3\,\text{Pa}}\right]$ | $\dfrac{\mathrm{d}\ln H_s^{cp}}{\mathrm{d}(1/T)}$<br><br>[K] | Reference | Type | Note |
|---|---|---|---|---|---|
| MCM:NBCKOOH | $4.8\times10^5$ | | Wang et al. (2017) | Q | 80, 238 |
| $C_{14}H_{23}NO_6$ | $4.4\times10^5$ | | Wang et al. (2017) | Q | 80, 239 |
| IMGMBHKHHVQTQK-UHFFFAOYSA-N | $1.5\times10^5$ | | Wang et al. (2017) | Q | 80, 240 |
| MCM:C141PAN | $1.9\times10^3$ | | Wang et al. (2017) | Q | 80, 238 |
| $C_{15}H_{23}NO_6$ | $1.8\times10^3$ | | Wang et al. (2017) | Q | 80, 239 |
| PGNAVHCNSKAOHZ-UHFFFAOYSA-N | $2.3\times10^1$ | | Wang et al. (2017) | Q | 80, 240 |
| MCM:C4PAN10 | $5.1\times10^7$ | | Wang et al. (2017) | Q | 80, 238 |
| $C_4H_5NO_8$ | $1.9\times10^7$ | | Wang et al. (2017) | Q | 80, 239 |
| JOAGXXIRMSWAAZ-UHFFFAOYSA-N | $2.4\times10^1$ | | Wang et al. (2017) | Q | 80, 240 |
| MCM:C4PAN6 | $8.1\times10^4$ | | Wang et al. (2017) | Q | 80, 238 |
| $C_4H_5NO_7$ | $5.4\times10^5$ | | Wang et al. (2017) | Q | 80, 239 |
| QRAIBPAZBCTEOB-UHFFFAOYSA-N | $2.0\times10^{-1}$ | | Wang et al. (2017) | Q | 80, 240 |
| MCM:CO3C4NO3OH | $8.9\times10^2$ | | Wang et al. (2017) | Q | 80, 238 |
| $C_4H_7NO_5$ | $3.2\times10^4$ | | Wang et al. (2017) | Q | 80, 239 |
| HZFDSYFRCPAXEA-UHFFFAOYSA-N | $2.5\times10^1$ | | Wang et al. (2017) | Q | 80, 240 |
| MCM:HMVKANO3 | $1.2\times10^3$ | | Wang et al. (2017) | Q | 80, 238 |
| $C_4H_7NO_5$ | $1.4\times10^4$ | | Wang et al. (2017) | Q | 80, 239 |
| UFDPWCOBIOWDKA-UHFFFAOYSA-N | $5.9$ | | Wang et al. (2017) | Q | 80, 240 |
| MCM:HMVKNO3 | $3.0\times10^6$ | | Wang et al. (2017) | Q | 80, 238 |
| $C_4H_7NO_6$ | $3.5\times10^7$ | | Wang et al. (2017) | Q | 80, 239 |
| NCLLRWDJHCXXDR-UHFFFAOYSA-N | $2.3\times10^3$ | | Wang et al. (2017) | Q | 80, 240 |
| MCM:HNBIACET | $5.9\times10^5$ | | Wang et al. (2017) | Q | 80, 238 |
| $C_4H_5NO_6$ | $1.4\times10^6$ | | Wang et al. (2017) | Q | 80, 239 |
| HPGBXXMJBUDRLY-UHFFFAOYSA-N | $4.7$ | | Wang et al. (2017) | Q | 80, 240 |
| MCM:HNMVKOH | $1.1\times10^6$ | | Wang et al. (2017) | Q | 80, 238 |
| $C_4H_7NO_6$ | $6.8\times10^6$ | | Wang et al. (2017) | Q | 80, 239 |
| KPZZNRQGHDTQNJ-UHFFFAOYSA-N | $1.1\times10^3$ | | Wang et al. (2017) | Q | 80, 240 |
| MCM:HNMVKOOH | $1.8\times10^9$ | | Wang et al. (2017) | Q | 80, 238 |
| $C_4H_7NO_7$ | $2.4\times10^7$ | | Wang et al. (2017) | Q | 80, 239 |
| CTKAFSDGNPEZFZ-UHFFFAOYSA-N | $1.0\times10^5$ | | Wang et al. (2017) | Q | 80, 240 |
| MCM:MVKNO3 | $2.4\times10^4$ | | Wang et al. (2017) | Q | 80, 238 |
| $C_4H_7NO_5$ | $2.0\times10^4$ | | Wang et al. (2017) | Q | 80, 239 |
| SCJOQBCZNBCTOQ-UHFFFAOYSA-N | $4.9\times10^1$ | | Wang et al. (2017) | Q | 80, 240 |
| MCM:MVKOHANO3 | $6.9\times10^5$ | | Wang et al. (2017) | Q | 80, 238 |
| $C_4H_7NO_6$ | $5.5\times10^5$ | | Wang et al. (2017) | Q | 80, 239 |
| DMSMZXCVKMBQGA-UHFFFAOYSA-N | $1.5\times10^3$ | | Wang et al. (2017) | Q | 80, 240 |
| MCM:C4MCONO3OH | $4.8\times10^2$ | | Wang et al. (2017) | Q | 80, 238 |
| $C_5H_9NO_5$ | $1.5\times10^4$ | | Wang et al. (2017) | Q | 80, 239 |
| VRAOJUVNSLNISW-UHFFFAOYSA-N | $1.5\times10^1$ | | Wang et al. (2017) | Q | 80, 240 |



Table A4.6: Nitrates ($RONO_2$) (... continued)

| Substance Formula (Trivial Name) [CAS Registry Number] InChIKey | $H_s^{cp}$ (at $T^\ominus$) $\left[ \dfrac{\text{mol}}{\text{m}^3\,\text{Pa}} \right]$ | $\dfrac{\text{d}\ln H_s^{cp}}{\text{d}(1/T)}$ [K] | Reference | Type | Note |
|---|---|---|---|---|---|
| MCM:C517NO3 | $1.6\times10^4$ | | Wang et al. (2017) | Q | 80, 238 |
| $C_5H_9NO_5$ | $1.9\times10^5$ | | Wang et al. (2017) | Q | 80, 239 |
| DLLRFAPEEPVGBN-UHFFFAOYSA-N | $7.8\times10^1$ | | Wang et al. (2017) | Q | 80, 240 |
| MCM:C51NO3 | $1.9\times10^4$ | | Wang et al. (2017) | Q | 80, 238 |
| $C_5H_9NO_5$ | $2.3\times10^5$ | | Wang et al. (2017) | Q | 80, 239 |
| OMTWKQRBUUWUQB-UHFFFAOYSA-N | $2.6\times10^2$ | | Wang et al. (2017) | Q | 80, 240 |
| MCM:C5NO3CO4OH | $1.6\times10^4$ | | Wang et al. (2017) | Q | 80, 238 |
| $C_5H_9NO_5$ | $3.1\times10^5$ | | Wang et al. (2017) | Q | 80, 239 |
| GEKIHWHZVPPEBT-UHFFFAOYSA-N | $9.6\times10^1$ | | Wang et al. (2017) | Q | 80, 240 |
| MCM:CO2OH3MPAN | $4.5\times10^4$ | | Wang et al. (2017) | Q | 80, 238 |
| $C_5H_7NO_7$ | $1.6\times10^5$ | | Wang et al. (2017) | Q | 80, 239 |
| KOFNDQLSBTYFBZ-UHFFFAOYSA-N | $6.6\times10^{-2}$ | | Wang et al. (2017) | Q | 80, 240 |
| MCM:CO3H4PAN | $6.5\times10^4$ | | Wang et al. (2017) | Q | 80, 238 |
| $C_5H_7NO_7$ | $2.8\times10^5$ | | Wang et al. (2017) | Q | 80, 239 |
| KVXLZPRANFUVNJ-UHFFFAOYSA-N | $1.6\times10^{-1}$ | | Wang et al. (2017) | Q | 80, 240 |
| MCM:H1C23C4PAN | $1.9\times10^8$ | | Wang et al. (2017) | Q | 80, 238 |
| $C_5H_5NO_8$ | $7.4\times10^7$ | | Wang et al. (2017) | Q | 80, 239 |
| ZNIKVNWPMOTSJK-UHFFFAOYSA-N | $2.2$ | | Wang et al. (2017) | Q | 80, 240 |
| MCM:H3C2C4PAN | $2.8\times10^5$ | | Wang et al. (2017) | Q | 80, 238 |
| $C_5H_7NO_7$ | $2.2\times10^6$ | | Wang et al. (2017) | Q | 80, 239 |
| UDPFGHOTTZCMEQ-UHFFFAOYSA-N | $4.1$ | | Wang et al. (2017) | Q | 80, 240 |
| MCM:HMVKBPAN | $5.5\times10^6$ | | Wang et al. (2017) | Q | 80, 238 |
| $C_5H_7NO_7$ | $3.6\times10^6$ | | Wang et al. (2017) | Q | 80, 239 |
| VVIGYLNPENSKGX-UHFFFAOYSA-N | $3.0$ | | Wang et al. (2017) | Q | 80, 240 |
| MCM:INANCO | $1.5\times10^6$ | | Wang et al. (2017) | Q | 80, 238 |
| $C_5H_8N_2O_8$ | $1.2\times10^6$ | | Wang et al. (2017) | Q | 80, 239 |
| IESHUNGXVKLDSS-UHFFFAOYSA-N | $1.5$ | | Wang et al. (2017) | Q | 80, 240 |
| MCM:INB1CO | $1.7\times10^6$ | | Wang et al. (2017) | Q | 80, 238 |
| $C_5H_9NO_6$ | $1.8\times10^7$ | | Wang et al. (2017) | Q | 80, 239 |
| ASROEBVFUUHBLI-UHFFFAOYSA-N | $2.1\times10^2$ | | Wang et al. (2017) | Q | 80, 240 |
| MCM:INCCO | $3.9\times10^5$ | | Wang et al. (2017) | Q | 80, 238 |
| $C_5H_9NO_6$ | $2.9\times10^5$ | | Wang et al. (2017) | Q | 80, 239 |
| HXWBEXHMXOXGKY-UHFFFAOYSA-N | $3.2\times10^2$ | | Wang et al. (2017) | Q | 80, 240 |
| MCM:C4COMOHPAN | $2.8\times10^7$ | | Wang et al. (2017) | Q | 80, 238 |
| $C_6H_7NO_8$ | $9.3\times10^6$ | | Wang et al. (2017) | Q | 80, 239 |
| KHOZKKAVCHSHMN-UHFFFAOYSA-N | $3.6\times10^{-2}$ | | Wang et al. (2017) | Q | 80, 240 |
| MCM:C4MOHOPAN | $6.0\times10^4$ | | Wang et al. (2017) | Q | 80, 238 |
| $C_6H_9NO_7$ | $1.9\times10^5$ | | Wang et al. (2017) | Q | 80, 239 |
| LVRCIQOIMYLYHB-UHFFFAOYSA-N | $3.8\times10^{-2}$ | | Wang et al. (2017) | Q | 80, 240 |





Table A4.6: Nitrates ($RONO_2$) (...continued)

| Substance Formula (Trivial Name) [CAS Registry Number] InChIKey | $H_s^{cp}$ (at $T^{\ominus}$) $\left[\dfrac{\text{mol}}{\text{m}^3\,\text{Pa}}\right]$ | $\dfrac{\text{d}\ln H_s^{cp}}{\text{d}(1/T)}$ [K] | Reference | Type | Note |
|---|---|---|---|---|---|
| MCM:C517PAN | $4.9\times10^6$ | | Wang et al. (2017) | Q | 80, 238 |
| $C_6H_9NO_7$ | $9.1\times10^6$ | | Wang et al. (2017) | Q | 80, 239 |
| BRUHFHISVNPPNB-UHFFFAOYSA-N | $4.0\times10^1$ | | Wang et al. (2017) | Q | 80, 240 |
| MCM:C519PAN | $4.9\times10^6$ | | Wang et al. (2017) | Q | 80, 238 |
| $C_6H_9NO_7$ | $9.6\times10^6$ | | Wang et al. (2017) | Q | 80, 239 |
| QFUOUYQJXYVTOO-UHFFFAOYSA-N | $3.6\times10^1$ | | Wang et al. (2017) | Q | 80, 240 |
| MCM:C5O45OHPAN | $5.6\times10^4$ | | Wang et al. (2017) | Q | 80, 238 |
| $C_6H_9NO_7$ | $1.7\times10^5$ | | Wang et al. (2017) | Q | 80, 239 |
| VCYALMFEDUSLKE-UHFFFAOYSA-N | $5.1\times10^{-2}$ | | Wang et al. (2017) | Q | 80, 240 |
| MCM:C610NO3 | $1.8\times10^4$ | | Wang et al. (2017) | Q | 80, 238 |
| $C_6H_{11}NO_5$ | $1.5\times10^5$ | | Wang et al. (2017) | Q | 80, 239 |
| KOUUCDJHLKRSEQ-UHFFFAOYSA-N | $2.5\times10^2$ | | Wang et al. (2017) | Q | 80, 240 |
| MCM:C614NO3 | $1.2\times10^7$ | | Wang et al. (2017) | Q | 80, 238 |
| $C_6H_9NO_6$ | $8.9\times10^6$ | | Wang et al. (2017) | Q | 80, 239 |
| QPXWFZAVXNBNFW-UHFFFAOYSA-N | $2.8\times10^2$ | | Wang et al. (2017) | Q | 80, 240 |
| MCM:C61NO3 | $1.8\times10^4$ | | Wang et al. (2017) | Q | 80, 238 |
| $C_6H_{11}NO_5$ | $1.4\times10^5$ | | Wang et al. (2017) | Q | 80, 239 |
| SYDWDJOZRNVKSU-UHFFFAOYSA-N | $6.6\times10^1$ | | Wang et al. (2017) | Q | 80, 240 |
| MCM:C63NO3 | $1.6\times10^4$ | | Wang et al. (2017) | Q | 80, 238 |
| $C_6H_{11}NO_5$ | $9.8\times10^4$ | | Wang et al. (2017) | Q | 80, 239 |
| RHTCZESJRHZQFA-UHFFFAOYSA-N | $3.1\times10^2$ | | Wang et al. (2017) | Q | 80, 240 |
| MCM:C64NO3 | $1.1\times10^4$ | | Wang et al. (2017) | Q | 80, 238 |
| $C_6H_{11}NO_5$ | $8.3\times10^4$ | | Wang et al. (2017) | Q | 80, 239 |
| IYYWGRLOLOMZAZ-UHFFFAOYSA-N | $1.3\times10^1$ | | Wang et al. (2017) | Q | 80, 240 |
| MCM:C6CONO34OH | $1.8\times10^4$ | | Wang et al. (2017) | Q | 80, 238 |
| $C_6H_{11}NO_5$ | $1.6\times10^4$ | | Wang et al. (2017) | Q | 80, 239 |
| OLTUAQAPCXKDOJ-UHFFFAOYSA-N | 2.7 | | Wang et al. (2017) | Q | 80, 240 |
| MCM:C6NO3CO4OH | $1.3\times10^4$ | | Wang et al. (2017) | Q | 80, 238 |
| $C_6H_{11}NO_5$ | $1.8\times10^5$ | | Wang et al. (2017) | Q | 80, 239 |
| WWUUDQUMERSNEM-UHFFFAOYSA-N | $4.8\times10^1$ | | Wang et al. (2017) | Q | 80, 240 |
| MCM:C6NO3CO5OH | $1.4\times10^4$ | | Wang et al. (2017) | Q | 80, 238 |
| $C_6H_{11}NO_5$ | $8.1\times10^4$ | | Wang et al. (2017) | Q | 80, 239 |
| IZBQRALUURUKJG-UHFFFAOYSA-N | $1.8\times10^1$ | | Wang et al. (2017) | Q | 80, 240 |
| MCM:C6PAN9 | $1.6\times10^5$ | | Wang et al. (2017) | Q | 80, 238 |
| $C_6H_9NO_7$ | $5.1\times10^5$ | | Wang et al. (2017) | Q | 80, 239 |
| NJFFIXOWLQJQIZ-UHFFFAOYSA-N | $6.8\times10^{-1}$ | | Wang et al. (2017) | Q | 80, 240 |
| MCM:MIBKAOHNO3 | $1.1\times10^4$ | | Wang et al. (2017) | Q | 80, 238 |
| $C_6H_{11}NO_5$ | $6.3\times10^4$ | | Wang et al. (2017) | Q | 80, 239 |
| YCHDRSSXCQCYSO-UHFFFAOYSA-N | $1.3\times10^1$ | | Wang et al. (2017) | Q | 80, 240 |





Table A4.6: Nitrates ($RONO_2$) (...continued)

| Substance Formula (Trivial Name) [CAS Registry Number] InChIKey | $H_s^{cp}$ (at $T^{\ominus}$) $\left[\dfrac{\mathrm{mol}}{\mathrm{m^3\,Pa}}\right]$ | $\dfrac{\mathrm{d}\ln H_s^{cp}}{\mathrm{d}(1/T)}$ [K] | Reference | Type | Note |
|---|---|---|---|---|---|
| MCM:C61CPAN $C_7H_9NO_8$ FWCVDVIZOKXUGJ-UHFFFAOYSA-N | $3.1\times10^7$ $1.2\times10^7$ $1.2\times10^{-1}$ | | Wang et al. (2017) Wang et al. (2017) Wang et al. (2017) | Q Q Q | 80, 238 80, 239 80, 240 |
| MCM:C62CPAN $C_7H_9NO_8$ LHOMRHYSYFWOPZ-UHFFFAOYSA-N | $3.6\times10^7$ $1.3\times10^7$ $7.4\times10^{-2}$ | | Wang et al. (2017) Wang et al. (2017) Wang et al. (2017) | Q Q Q | 80, 238 80, 239 80, 240 |
| MCM:C712NO3 $C_7H_{13}NO_5$ RPPGGLBPWJPSSV-UHFFFAOYSA-N | $1.0\times10^4$ $5.4\times10^4$ $7.1$ | | Wang et al. (2017) Wang et al. (2017) Wang et al. (2017) | Q Q Q | 80, 238 80, 239 80, 240 |
| MCM:C719NO3 $C_7H_{11}NO_6$ IXGSPJJVHFTXPK-UHFFFAOYSA-N | $1.0\times10^8$ $4.8\times10^9$ $2.8\times10^6$ | | Wang et al. (2017) Wang et al. (2017) Wang et al. (2017) | Q Q Q | 80, 238 80, 239 80, 240 |
| MCM:C72NO3 $C_7H_{13}NO_5$ MWIGFUUCEWIWBB-UHFFFAOYSA-N | $1.6\times10^4$ $6.5\times10^4$ $2.6\times10^1$ | | Wang et al. (2017) Wang et al. (2017) Wang et al. (2017) | Q Q Q | 80, 238 80, 239 80, 240 |
| MCM:C77NO3 $C_7H_{13}NO_5$ KZCKPUAVFWISTB-UHFFFAOYSA-N | $1.0\times10^4$ $5.0\times10^4$ $7.4\times10^1$ | | Wang et al. (2017) Wang et al. (2017) Wang et al. (2017) | Q Q Q | 80, 238 80, 239 80, 240 |
| MCM:H3C25C6PAN $C_7H_9NO_8$ GUUOCMMOMFUPSI-UHFFFAOYSA-N | $1.4\times10^8$ $6.8\times10^8$ $1.6\times10^2$ | | Wang et al. (2017) Wang et al. (2017) Wang et al. (2017) | Q Q Q | 80, 238 80, 239 80, 240 |
| MCM:C6MOHCOPAN $C_8H_9NO_8$ PDGRPGJWHYLYKK-UHFFFAOYSA-N | $9.6\times10^7$ $1.1\times10^9$ $6.2\times10^{-1}$ | | Wang et al. (2017) Wang et al. (2017) Wang et al. (2017) | Q Q Q | 80, 238 80, 239 80, 240 |
| MCM:C7CO2OHPAN $C_8H_9NO_8$ MICPXOSCTLYWLE-UHFFFAOYSA-N | $9.6\times10^7$ $1.1\times10^9$ $4.9\times10^{-1}$ | | Wang et al. (2017) Wang et al. (2017) Wang et al. (2017) | Q Q Q | 80, 238 80, 239 80, 240 |
| MCM:C829NO3 $C_8H_{13}NO_6$ CDMYSTRAPZWSHD-UHFFFAOYSA-N | $4.2\times10^6$ $1.3\times10^6$ $2.5\times10^2$ | | Wang et al. (2017) Wang et al. (2017) Wang et al. (2017) | Q Q Q | 80, 238 80, 239 80, 240 |
| MCM:C82NO3 $C_8H_{15}NO_5$ HUKHBWOLGKLCRF-UHFFFAOYSA-N | $1.3\times10^4$ $4.1\times10^4$ $7.4$ | | Wang et al. (2017) Wang et al. (2017) Wang et al. (2017) | Q Q Q | 80, 238 80, 239 80, 240 |
| MCM:C6EO2OHPAN $C_9H_{11}NO_8$ YGQCKHJQVYPZEV-UHFFFAOYSA-N | $7.6\times10^7$ $6.9\times10^8$ $3.2\times10^{-1}$ | | Wang et al. (2017) Wang et al. (2017) Wang et al. (2017) | Q Q Q | 80, 238 80, 239 80, 240 |
| MCM:C7MJPPAN $C_9H_{11}NO_8$ XJDHGIQDEMCTOX-UHFFFAOYSA-N | $7.6\times10^7$ $6.5\times10^8$ $8.9\times10^{-1}$ | | Wang et al. (2017) Wang et al. (2017) Wang et al. (2017) | Q Q Q | 80, 238 80, 239 80, 240 |





Table A4.6: Nitrates ($RONO_2$) (... continued)

| Substance Formula (Trivial Name) [CAS Registry Number] InChIKey | $H_s^{cp}$ (at $T^\ominus$) $\left[\dfrac{\text{mol}}{\text{m}^3\,\text{Pa}}\right]$ | $\dfrac{\text{d}\ln H_s^{cp}}{\text{d}(1/T)}$ [K] | Reference | Type | Note |
|---|---|---|---|---|---|
| MCM:C7MOHCOPAN | $5.5\times10^7$ | | Wang et al. (2017) | Q | 80, 238 |
| $C_9H_{11}NO_8$ | $1.7\times10^9$ | | Wang et al. (2017) | Q | 80, 239 |
| SORJSZWEJQTGJA-UHFFFAOYSA-N | 1.3 | | Wang et al. (2017) | Q | 80, 240 |
| MCM:C927NO3 | $1.0\times10^4$ | | Wang et al. (2017) | Q | 80, 238 |
| $C_9H_{15}NO_5$ | $1.1\times10^5$ | | Wang et al. (2017) | Q | 80, 239 |
| RKOGSSYMWVPWJW-UHFFFAOYSA-N | $2.9\times10^3$ | | Wang et al. (2017) | Q | 80, 240 |
| MCM:C93NO3 | $1.0\times10^4$ | | Wang et al. (2017) | Q | 80, 238 |
| $C_9H_{17}NO_5$ | $3.4\times10^4$ | | Wang et al. (2017) | Q | 80, 239 |
| OEEOAFFOONSSLD-UHFFFAOYSA-N | 5.1 | | Wang et al. (2017) | Q | 80, 240 |
| MCM:C98NO3 | $4.0\times10^6$ | | Wang et al. (2017) | Q | 80, 238 |
| $C_9H_{15}NO_6$ | $5.6\times10^6$ | | Wang et al. (2017) | Q | 80, 239 |
| PQCIVFJRBSNULM-UHFFFAOYSA-N | $1.8\times10^3$ | | Wang et al. (2017) | Q | 80, 240 |
| MCM:LMKANO3 | $2.3\times10^4$ | | Wang et al. (2017) | Q | 80, 238 |
| $C_9H_{15}NO_5$ | $2.2\times10^5$ | | Wang et al. (2017) | Q | 80, 239 |
| OCTKYXAOGSBZDD-UHFFFAOYSA-N | $1.3\times10^4$ | | Wang et al. (2017) | Q | 80, 240 |
| MCM:LMKBNO3 | $2.3\times10^4$ | | Wang et al. (2017) | Q | 80, 238 |
| $C_9H_{15}NO_5$ | $2.6\times10^5$ | | Wang et al. (2017) | Q | 80, 239 |
| MVWAQQXSBOJSHS-UHFFFAOYSA-N | $2.1\times10^3$ | | Wang et al. (2017) | Q | 80, 240 |
| MCM:C103NO3 | $8.3\times10^3$ | | Wang et al. (2017) | Q | 80, 238 |
| $C_{10}H_{19}NO_5$ | $3.0\times10^4$ | | Wang et al. (2017) | Q | 80, 239 |
| NTZGBOUJHFKYIP-UHFFFAOYSA-N | 4.8 | | Wang et al. (2017) | Q | 80, 240 |
| MCM:C920PAN | $3.1\times10^5$ | | Wang et al. (2017) | Q | 80, 238 |
| $C_{10}H_{15}NO_7$ | $5.9\times10^5$ | | Wang et al. (2017) | Q | 80, 239 |
| JPFVAVYDZBUWDH-UHFFFAOYSA-N | $1.9\times10^2$ | | Wang et al. (2017) | Q | 80, 240 |
| MCM:C113NO3 | $7.4\times10^3$ | | Wang et al. (2017) | Q | 80, 238 |
| $C_{11}H_{21}NO_5$ | $2.8\times10^4$ | | Wang et al. (2017) | Q | 80, 239 |
| XIMKJTRLGBXFHP-UHFFFAOYSA-N | 9.3 | | Wang et al. (2017) | Q | 80, 240 |
| MCM:C123NO3 | $6.0\times10^3$ | | Wang et al. (2017) | Q | 80, 238 |
| $C_{12}H_{23}NO_5$ | $2.5\times10^4$ | | Wang et al. (2017) | Q | 80, 239 |
| MVMJPHZGIRWSLB-UHFFFAOYSA-N | 3.4 | | Wang et al. (2017) | Q | 80, 240 |
| MCM:C133NO3 | $1.2\times10^9$ | | Wang et al. (2017) | Q | 80, 238 |
| $C_{13}H_{21}NO_7$ | $5.1\times10^8$ | | Wang et al. (2017) | Q | 80, 239 |
| ODJJFBKGWDLDSF-UHFFFAOYSA-N | $1.8\times10^5$ | | Wang et al. (2017) | Q | 80, 240 |
| MCM:BCKANO3 | $2.5\times10^4$ | | Wang et al. (2017) | Q | 80, 238 |
| $C_{14}H_{23}NO_5$ | $6.0\times10^5$ | | Wang et al. (2017) | Q | 80, 239 |
| YULYIMMPXZFYHU-UHFFFAOYSA-N | $1.8\times10^4$ | | Wang et al. (2017) | Q | 80, 240 |
| MCM:BCKBNO3 | $2.5\times10^4$ | | Wang et al. (2017) | Q | 80, 238 |
| $C_{14}H_{23}NO_5$ | $1.0\times10^6$ | | Wang et al. (2017) | Q | 80, 239 |
| VHSFPWDPGIOBJY-UHFFFAOYSA-N | $1.9\times10^4$ | | Wang et al. (2017) | Q | 80, 240 |





Table A4.6: Nitrates ($RONO_2$) (. . . continued)

| Substance / Formula / (Trivial Name) / [CAS Registry Number] / InChIKey | $H_s^{cp}$ (at $T^\ominus$) $\left[\dfrac{\mathrm{mol}}{\mathrm{m^3\,Pa}}\right]$ | $\dfrac{\mathrm{d}\ln H_s^{cp}}{\mathrm{d}(1/T)}$ [K] | Reference | Type | Note |
|---|---|---|---|---|---|
| MCM:C142NO3 | $1.1\times10^4$ | | Wang et al. (2017) | Q | 80, 238 |
| $C_{14}H_{23}NO_5$ | $1.9\times10^5$ | | Wang et al. (2017) | Q | 80, 239 |
| AZGAJOUTVTYWPD-UHFFFAOYSA-N | $1.4\times10^4$ | | Wang et al. (2017) | Q | 80, 240 |
| MCM:C143NO3 | $3.4\times10^6$ | | Wang et al. (2017) | Q | 80, 238 |
| $C_{14}H_{23}NO_6$ | $5.0\times10^7$ | | Wang et al. (2017) | Q | 80, 239 |
| NNHMCYUMVAAOGX-UHFFFAOYSA-N | $1.0\times10^5$ | | Wang et al. (2017) | Q | 80, 240 |
| MCM:NMGLYOX | $6.0\times10^3$ | | Wang et al. (2017) | Q | 80, 238 |
| $C_3H_3NO_5$ | $4.1\times10^3$ | | Wang et al. (2017) | Q | 80, 239 |
| OPXOPVKLHYDIMX-UHFFFAOYSA-N | $3.4\times10^{-2}$ | | Wang et al. (2017) | Q | 80, 240 |
| MCM:C312COPAN | $2.2\times10^6$ | | Wang et al. (2017) | Q | 80, 238 |
| $C_4H_3NO_7$ | $2.8\times10^5$ | | Wang et al. (2017) | Q | 80, 239 |
| RRBQALAYWSLXKO-UHFFFAOYSA-N | $3.3\times10^{-2}$ | | Wang et al. (2017) | Q | 80, 240 |
| MCM:C4CONO3CO | $6.5\times10^3$ | | Wang et al. (2017) | Q | 80, 238 |
| $C_4H_5NO_5$ | $8.3\times10^3$ | | Wang et al. (2017) | Q | 80, 239 |
| RXLGSPGPVVHCFU-UHFFFAOYSA-N | $2.2\times10^{-1}$ | | Wang et al. (2017) | Q | 80, 240 |
| MCM:NCO23CHO | $3.6\times10^6$ | | Wang et al. (2017) | Q | 80, 238 |
| $C_4H_3NO_6$ | $9.8\times10^5$ | | Wang et al. (2017) | Q | 80, 239 |
| VZDGBDWMYVQZKU-UHFFFAOYSA-N | $1.1\times10^{-1}$ | | Wang et al. (2017) | Q | 80, 240 |
| MCM:C4CO2DBPAN | $6.9\times10^6$ | | Wang et al. (2017) | Q | 80, 238 |
| $C_5H_3NO_7$ | $2.5\times10^6$ | | Wang et al. (2017) | Q | 80, 239 |
| BTFAHSUMXINULH-UHFFFAOYSA-N | $4.9\times10^{-3}$ | | Wang et al. (2017) | Q | 80, 240 |
| MCM:C4M22CONO3 | $3.5\times10^3$ | | Wang et al. (2017) | Q | 80, 238 |
| $C_5H_7NO_5$ | $1.6\times10^3$ | | Wang et al. (2017) | Q | 80, 239 |
| FQGALMJVSZOCNN-UHFFFAOYSA-N | $5.1\times10^{-2}$ | | Wang et al. (2017) | Q | 80, 240 |
| MCM:C512NO3 | $4.4\times10^3$ | | Wang et al. (2017) | Q | 80, 238 |
| $C_5H_7NO_5$ | $2.0\times10^4$ | | Wang et al. (2017) | Q | 80, 239 |
| OAONLGDTCNNTSO-UHFFFAOYSA-N | $8.1\times10^1$ | | Wang et al. (2017) | Q | 80, 240 |
| MCM:CHOC3COPAN | $1.7\times10^6$ | | Wang et al. (2017) | Q | 80, 238 |
| $C_5H_5NO_7$ | $3.8\times10^5$ | | Wang et al. (2017) | Q | 80, 239 |
| MEMYBUIJIAQZMF-UHFFFAOYSA-N | $9.6\times10^{-1}$ | | Wang et al. (2017) | Q | 80, 240 |
| MCM:INANCOCHO | $4.0\times10^5$ | | Wang et al. (2017) | Q | 80, 238 |
| $C_5H_6N_2O_8$ | $2.6\times10^5$ | | Wang et al. (2017) | Q | 80, 239 |
| ALBXLNVOOLMBSV-UHFFFAOYSA-N | $1.5\times10^{-2}$ | | Wang et al. (2017) | Q | 80, 240 |
| MCM:C511PAN | $1.6\times10^6$ | | Wang et al. (2017) | Q | 80, 238 |
| $C_6H_7NO_7$ | $8.9\times10^5$ | | Wang et al. (2017) | Q | 80, 239 |
| ADLPISUKTIUEBV-UHFFFAOYSA-N | $8.3\times10^{-1}$ | | Wang et al. (2017) | Q | 80, 240 |
| MCM:C5124COPAN | $1.0\times10^9$ | | Wang et al. (2017) | Q | 80, 238 |
| $C_6H_5NO_8$ | $1.3\times10^7$ | | Wang et al. (2017) | Q | 80, 239 |
| DHCQJQARWDKNBK-UHFFFAOYSA-N | $2.1\times10^{-1}$ | | Wang et al. (2017) | Q | 80, 240 |





Table A4.6: Nitrates ($RONO_2$) (...continued)

| Substance Formula (Trivial Name) [CAS Registry Number] InChIKey | $H_s^{cp}$ (at $T^\ominus$) $\left[\dfrac{\text{mol}}{\text{m}^3\,\text{Pa}}\right]$ | $\dfrac{\text{d}\ln H_s^{cp}}{\text{d}(1/T)}$ [K] | Reference | Type | Note |
|---|---|---|---|---|---|
| MCM:C512PAN | $1.4\times10^6$ | | Wang et al. (2017) | Q | 80, 238 |
| $C_6H_7NO_7$ | $1.1\times10^6$ | | Wang et al. (2017) | Q | 80, 239 |
| OLSCQSOEJBIJSE-UHFFFAOYSA-N | $1.4\times10^1$ | | Wang et al. (2017) | Q | 80, 240 |
| MCM:C515PAN | $1.0\times10^9$ | | Wang et al. (2017) | Q | 80, 238 |
| $C_6H_5NO_8$ | $3.0\times10^7$ | | Wang et al. (2017) | Q | 80, 239 |
| QIUJGMPSWRWOQB-UHFFFAOYSA-N | 1.9 | | Wang et al. (2017) | Q | 80, 240 |
| MCM:C5CO2DBPAN | $4.7\times10^6$ | | Wang et al. (2017) | Q | 80, 238 |
| $C_6H_5NO_7$ | $2.1\times10^6$ | | Wang et al. (2017) | Q | 80, 239 |
| LCULQJDFGZNNJC-UHFFFAOYSA-N | $2.8\times10^{-3}$ | | Wang et al. (2017) | Q | 80, 240 |
| MCM:C5DBCO2PAN | $4.7\times10^6$ | | Wang et al. (2017) | Q | 80, 238 |
| $C_6H_5NO_7$ | $2.1\times10^6$ | | Wang et al. (2017) | Q | 80, 239 |
| QMYFWHUFARKUQS-UHFFFAOYSA-N | $3.6\times10^{-3}$ | | Wang et al. (2017) | Q | 80, 240 |
| MCM:C626NO3 | $4.0\times10^3$ | | Wang et al. (2017) | Q | 80, 238 |
| $C_6H_9NO_5$ | $5.1\times10^4$ | | Wang et al. (2017) | Q | 80, 239 |
| OCVBVQJPDKPHST-UHFFFAOYSA-N | $1.8\times10^1$ | | Wang et al. (2017) | Q | 80, 240 |
| MCM:C6COCHOPAN | $1.4\times10^6$ | | Wang et al. (2017) | Q | 80, 238 |
| $C_6H_7NO_7$ | $2.6\times10^6$ | | Wang et al. (2017) | Q | 80, 239 |
| YGLXJUWEIHLTAS-UHFFFAOYSA-N | $8.9\times10^1$ | | Wang et al. (2017) | Q | 80, 240 |
| MCM:C4DBM2PAN | $3.0\times10^6$ | | Wang et al. (2017) | Q | 80, 238 |
| $C_7H_7NO_7$ | $3.0\times10^6$ | | Wang et al. (2017) | Q | 80, 239 |
| LSIPSXVETZQMNK-UHFFFAOYSA-N | $5.4\times10^{-3}$ | | Wang et al. (2017) | Q | 80, 240 |
| MCM:C617PAN | $8.7\times10^5$ | | Wang et al. (2017) | Q | 80, 238 |
| $C_7H_9NO_7$ | $9.6\times10^4$ | | Wang et al. (2017) | Q | 80, 239 |
| PZZHENCPCXAAMZ-UHFFFAOYSA-N | $6.3\times10^{-1}$ | | Wang et al. (2017) | Q | 80, 240 |
| MCM:C618PAN | $8.7\times10^5$ | | Wang et al. (2017) | Q | 80, 238 |
| $C_7H_9NO_7$ | $7.8\times10^4$ | | Wang et al. (2017) | Q | 80, 239 |
| TUYXNXVHKYYCNT-UHFFFAOYSA-N | $5.6\times10^{-1}$ | | Wang et al. (2017) | Q | 80, 240 |
| MCM:C626PAN | $1.3\times10^6$ | | Wang et al. (2017) | Q | 80, 238 |
| $C_7H_9NO_7$ | $2.0\times10^6$ | | Wang et al. (2017) | Q | 80, 239 |
| JTOZIQOFBRFINW-UHFFFAOYSA-N | $1.6\times10^1$ | | Wang et al. (2017) | Q | 80, 240 |
| MCM:C717NO3 | $2.4\times10^6$ | | Wang et al. (2017) | Q | 80, 238 |
| $C_7H_9NO_6$ | $8.0\times10^5$ | | Wang et al. (2017) | Q | 80, 239 |
| YBNFZHXBUOLIKA-UHFFFAOYSA-N | $2.2\times10^1$ | | Wang et al. (2017) | Q | 80, 240 |
| MCM:C718NO3 | $2.2\times10^3$ | | Wang et al. (2017) | Q | 80, 238 |
| $C_7H_{11}NO_5$ | $3.5\times10^3$ | | Wang et al. (2017) | Q | 80, 239 |
| WMUFPKACUPNARK-UHFFFAOYSA-N | 3.8 | | Wang et al. (2017) | Q | 80, 240 |
| MCM:C731NO3 | $3.2\times10^3$ | | Wang et al. (2017) | Q | 80, 238 |
| $C_7H_{11}NO_5$ | $4.5\times10^4$ | | Wang et al. (2017) | Q | 80, 239 |
| WJSMASANGLXUCK-UHFFFAOYSA-N | $1.7\times10^2$ | | Wang et al. (2017) | Q | 80, 240 |





Table A4.6: Nitrates ($RONO_2$) (... continued)

| Substance<br>Formula<br>(Trivial Name)<br>[CAS Registry Number]<br>InChIKey | $H_s^{cp}$<br>(at $T^\ominus$)<br>$\left[\dfrac{\text{mol}}{\text{m}^3\,\text{Pa}}\right]$ | $\dfrac{\text{d}\ln H_s^{cp}}{\text{d}(1/T)}$<br><br>[K] | Reference | Type | Note |
|---|---|---|---|---|---|
| MCM:C7CO2DBPAN<br>$C_7H_7NO_7$<br>FJSOTAPJABBDID-UHFFFAOYSA-N | $3.7\times10^6$<br>$1.2\times10^6$<br>$2.6\times10^{-3}$ | | Wang et al. (2017)<br>Wang et al. (2017)<br>Wang et al. (2017) | Q<br>Q<br>Q | 80, 238<br>80, 239<br>80, 240 |
| MCM:NC61CO3H<br>$C_7H_7NO_9$<br>RCADYDMYBSYTRH-UHFFFAOYSA-N | $1.7\times10^{12}$<br>$6.2\times10^8$<br>$6.9\times10^1$ | | Wang et al. (2017)<br>Wang et al. (2017)<br>Wang et al. (2017) | Q<br>Q<br>Q | 80, 238<br>80, 239<br>80, 240 |
| MCM:NC6PAN1<br>$C_7H_6N_2O_{11}$<br>LNWBMFOWVOYBBT-UHFFFAOYSA-N | $6.6\times10^{10}$<br>$2.4\times10^8$<br>$2.0\times10^{-2}$ | | Wang et al. (2017)<br>Wang et al. (2017)<br>Wang et al. (2017) | Q<br>Q<br>Q | 80, 238<br>80, 239<br>80, 240 |
| MCM:C4DBMEPAN<br>$C_8H_9NO_7$<br>JDCDQJWNYIFIQT-UHFFFAOYSA-N | $2.5\times10^6$<br>$1.6\times10^6$<br>$3.5\times10^{-3}$ | | Wang et al. (2017)<br>Wang et al. (2017)<br>Wang et al. (2017) | Q<br>Q<br>Q | 80, 238<br>80, 239<br>80, 240 |
| MCM:C718PAN<br>$C_8H_{11}NO_7$<br>IZJQRCOOVGKQNV-UHFFFAOYSA-N | $7.1\times10^5$<br>$2.3\times10^5$<br>$1.4$ | | Wang et al. (2017)<br>Wang et al. (2017)<br>Wang et al. (2017) | Q<br>Q<br>Q | 80, 238<br>80, 239<br>80, 240 |
| MCM:C731PAN<br>$C_8H_{11}NO_7$<br>BCDBCVPVNWUZKV-UHFFFAOYSA-N | $1.1\times10^6$<br>$1.7\times10^6$<br>$4.4\times10^1$ | | Wang et al. (2017)<br>Wang et al. (2017)<br>Wang et al. (2017) | Q<br>Q<br>Q | 80, 238<br>80, 239<br>80, 240 |
| MCM:C8CO2DBPAN<br>$C_8H_9NO_7$<br>IPAWZRQBQOFPJA-UHFFFAOYSA-N | $3.4\times10^6$<br>$6.9\times10^5$<br>$1.4\times10^{-3}$ | | Wang et al. (2017)<br>Wang et al. (2017)<br>Wang et al. (2017) | Q<br>Q<br>Q | 80, 238<br>80, 239<br>80, 240 |
| MCM:C8DBCO2PAN<br>$C_8H_9NO_7$<br>ZDCAPCKMKJMPHV-UHFFFAOYSA-N | $3.5\times10^6$<br>$8.0\times10^5$<br>$1.0\times10^{-3}$ | | Wang et al. (2017)<br>Wang et al. (2017)<br>Wang et al. (2017) | Q<br>Q<br>Q | 80, 238<br>80, 239<br>80, 240 |
| MCM:C87PAN<br>$C_9H_{11}NO_8$<br>KZPDHKTVZKNFJF-UHFFFAOYSA-N | $6.0\times10^8$<br>$2.8\times10^7$<br>$7.8$ | | Wang et al. (2017)<br>Wang et al. (2017)<br>Wang et al. (2017) | Q<br>Q<br>Q | 80, 238<br>80, 239<br>80, 240 |
| MCM:C915NO3<br>$C_9H_{13}NO_5$<br>CFOCUCLSGKHNEG-UHFFFAOYSA-N | $5.9\times10^3$<br>$3.2\times10^4$<br>$5.6\times10^1$ | | Wang et al. (2017)<br>Wang et al. (2017)<br>Wang et al. (2017) | Q<br>Q<br>Q | 80, 238<br>80, 239<br>80, 240 |
| MCM:C916NO3<br>$C_9H_{13}NO_6$<br>VOXWFVUGDBUBCW-UHFFFAOYSA-N | $1.5\times10^6$<br>$1.1\times10^6$<br>$4.9\times10^1$ | | Wang et al. (2017)<br>Wang et al. (2017)<br>Wang et al. (2017) | Q<br>Q<br>Q | 80, 238<br>80, 239<br>80, 240 |
| MCM:C918NO3<br>$C_9H_{13}NO_5$<br>AMOFMDWAGAYJCS-UHFFFAOYSA-N | $5.9\times10^3$<br>$5.6\times10^3$<br>$5.6\times10^1$ | | Wang et al. (2017)<br>Wang et al. (2017)<br>Wang et al. (2017) | Q<br>Q<br>Q | 80, 238<br>80, 239<br>80, 240 |
| MCM:C919NO3<br>$C_9H_{13}NO_6$<br>DCZHYWWXGROBKB-UHFFFAOYSA-N | $1.5\times10^6$<br>$1.0\times10^6$<br>$8.0\times10^2$ | | Wang et al. (2017)<br>Wang et al. (2017)<br>Wang et al. (2017) | Q<br>Q<br>Q | 80, 238<br>80, 239<br>80, 240 |





Table A4.6: Nitrates ($RONO_2$) (... continued)

| Substance Formula (Trivial Name) [CAS Registry Number] InChIKey | $H_s^{cp}$ (at $T^{\ominus}$) $\left[\dfrac{\mathrm{mol}}{\mathrm{m^3\,Pa}}\right]$ | $\dfrac{\mathrm{d}\ln H_s^{cp}}{\mathrm{d}(1/T)}$ [K] | Reference | Type | Note |
|---|---|---|---|---|---|
| MCM:C930NO3 | $1.1\times10^6$ | | Wang et al. (2017) | Q | 80, 238 |
| $C_9H_{13}NO_6$ | $5.6\times10^5$ | | Wang et al. (2017) | Q | 80, 239 |
| DDSHYSHFPVKRPI-UHFFFAOYSA-N | $1.8\times10^1$ | | Wang et al. (2017) | Q | 80, 240 |
| MCM:C1010NO3 | $3.0\times10^3$ | | Wang et al. (2017) | Q | 80, 238 |
| $C_{10}H_{15}NO_5$ | $8.5\times10^3$ | | Wang et al. (2017) | Q | 80, 239 |
| GVXIDPZAUKXFPR-UHFFFAOYSA-N | $4.0\times10^1$ | | Wang et al. (2017) | Q | 80, 240 |
| MCM:C1012NO3 | $1.5\times10^3$ | | Wang et al. (2017) | Q | 80, 238 |
| $C_{10}H_{17}NO_5$ | $8.1\times10^3$ | | Wang et al. (2017) | Q | 80, 239 |
| XKIZNDJDYSDNBL-UHFFFAOYSA-N | $2.7\times10^2$ | | Wang et al. (2017) | Q | 80, 240 |
| MCM:C106NO3 | $1.0\times10^6$ | | Wang et al. (2017) | Q | 80, 238 |
| $C_{10}H_{15}NO_6$ | $7.3\times10^5$ | | Wang et al. (2017) | Q | 80, 239 |
| DSEFLMWIAZKUSJ-UHFFFAOYSA-N | $4.9\times10^2$ | | Wang et al. (2017) | Q | 80, 240 |
| MCM:C108NO3 | $1.0\times10^6$ | | Wang et al. (2017) | Q | 80, 238 |
| $C_{10}H_{15}NO_6$ | $2.2\times10^5$ | | Wang et al. (2017) | Q | 80, 239 |
| MVSLQVXNIWKHQS-UHFFFAOYSA-N | $6.6\times10^1$ | | Wang et al. (2017) | Q | 80, 240 |
| MCM:NLIMALOOH | $1.1\times10^8$ | | Wang et al. (2017) | Q | 80, 238 |
| $C_{10}H_{17}NO_7$ | $3.4\times10^7$ | | Wang et al. (2017) | Q | 80, 239 |
| NTLQQKIUYMIZPN-UHFFFAOYSA-N | $2.8\times10^5$ | | Wang et al. (2017) | Q | 80, 240 |
| MCM:PINALNO3 | $3.4\times10^3$ | | Wang et al. (2017) | Q | 80, 238 |
| $C_{10}H_{15}NO_5$ | $1.6\times10^4$ | | Wang et al. (2017) | Q | 80, 239 |
| FNTMCASBPWQXGD-UHFFFAOYSA-N | $3.8\times10^2$ | | Wang et al. (2017) | Q | 80, 240 |
| MCM:C116NO3 | $4.3\times10^3$ | | Wang et al. (2017) | Q | 80, 238 |
| $C_{11}H_{17}NO_5$ | $3.7\times10^4$ | | Wang et al. (2017) | Q | 80, 239 |
| AWSJYNTZBDMUKU-UHFFFAOYSA-N | $3.8\times10^1$ | | Wang et al. (2017) | Q | 80, 240 |
| MCM:C116PAN | $1.4\times10^6$ | | Wang et al. (2017) | Q | 80, 238 |
| $C_{12}H_{17}NO_7$ | $2.3\times10^6$ | | Wang et al. (2017) | Q | 80, 239 |
| IHJLCJPVFFZKMO-UHFFFAOYSA-N | $1.2\times10^1$ | | Wang et al. (2017) | Q | 80, 240 |
| MCM:C1210NO3 | $3.4\times10^3$ | | Wang et al. (2017) | Q | 80, 238 |
| $C_{12}H_{19}NO_5$ | $4.1\times10^4$ | | Wang et al. (2017) | Q | 80, 239 |
| IJHSXQWUSMROBJ-UHFFFAOYSA-N | $7.4\times10^1$ | | Wang et al. (2017) | Q | 80, 240 |
| MCM:C1210PAN | $1.1\times10^6$ | | Wang et al. (2017) | Q | 80, 238 |
| $C_{13}H_{19}NO_7$ | $2.0\times10^6$ | | Wang et al. (2017) | Q | 80, 239 |
| PWZZEUPXUFAYML-UHFFFAOYSA-N | $1.0\times10^2$ | | Wang et al. (2017) | Q | 80, 240 |
| MCM:NBCALOOH | $1.1\times10^8$ | | Wang et al. (2017) | Q | 80, 238 |
| $C_{15}H_{25}NO_7$ | $8.5\times10^7$ | | Wang et al. (2017) | Q | 80, 239 |
| TUYLAULXHOKFAG-UHFFFAOYSA-N | $7.3\times10^3$ | | Wang et al. (2017) | Q | 80, 240 |
| MCM:H3NCO2CHO | $1.8\times10^5$ | | Wang et al. (2017) | Q | 80, 238 |
| $C_4H_5NO_6$ | $3.6\times10^6$ | | Wang et al. (2017) | Q | 80, 239 |
| VCTGKRYJUDQYFK-UHFFFAOYSA-N | $3.4$ | | Wang et al. (2017) | Q | 80, 240 |





Table A4.6: Nitrates ($RONO_2$) (...continued)

| Substance Formula (Trivial Name) [CAS Registry Number] InChIKey | $H_s^{cp}$ (at $T^\ominus$) $\left[\dfrac{\mathrm{mol}}{\mathrm{m^3\,Pa}}\right]$ | $\dfrac{\mathrm{d}\ln H_s^{cp}}{\mathrm{d}(1/T)}$ [K] | Reference | Type | Note |
|---|---|---|---|---|---|
| MCM:HMVKNGLYOX | $2.2\times10^7$ | | Wang et al. (2017) | Q | 80, 238 |
| $C_4H_5NO_6$ | $9.3\times10^5$ | | Wang et al. (2017) | Q | 80, 239 |
| VDDXFUNTPSDTJP-UHFFFAOYSA-N | 9.3 | | Wang et al. (2017) | Q | 80, 240 |
| MCM:INB1GLYOX | $1.2\times10^7$ | | Wang et al. (2017) | Q | 80, 238 |
| $C_5H_7NO_6$ | $3.1\times10^5$ | | Wang et al. (2017) | Q | 80, 239 |
| NAZCWKUVRXJZPX-UHFFFAOYSA-N | 3.0 | | Wang et al. (2017) | Q | 80, 240 |
| MCM:INCGLYOX | $6.2\times10^5$ | | Wang et al. (2017) | Q | 80, 238 |
| $C_5H_7NO_6$ | $4.8\times10^5$ | | Wang et al. (2017) | Q | 80, 239 |
| DGCXGPGOOTUCCO-UHFFFAOYSA-N | $3.0\times10^{-1}$ | | Wang et al. (2017) | Q | 80, 240 |
| MCM:C5CO2OHPAN | $2.1\times10^8$ | | Wang et al. (2017) | Q | 80, 238 |
| $C_6H_5NO_8$ | $1.7\times10^9$ | | Wang et al. (2017) | Q | 80, 239 |
| MLSWEUDOYVEBFO-UHFFFAOYSA-N | $7.6\times10^{-1}$ | | Wang et al. (2017) | Q | 80, 240 |
| MCM:C6CO2OHPAN | $1.4\times10^8$ | | Wang et al. (2017) | Q | 80, 238 |
| $C_7H_7NO_8$ | $1.4\times10^9$ | | Wang et al. (2017) | Q | 80, 239 |
| GITGCCFTTYJWBT-UHFFFAOYSA-N | $4.2\times10^{-1}$ | | Wang et al. (2017) | Q | 80, 240 |
| MCM:C5M2OHOPAN | $9.6\times10^7$ | | Wang et al. (2017) | Q | 80, 238 |
| $C_8H_9NO_8$ | $2.0\times10^9$ | | Wang et al. (2017) | Q | 80, 239 |
| MJYXUWCXTAYXTN-UHFFFAOYSA-N | $7.3\times10^{-1}$ | | Wang et al. (2017) | Q | 80, 240 |
| MCM:C7OHCO2PAN | $1.1\times10^8$ | | Wang et al. (2017) | Q | 80, 238 |
| $C_8H_9NO_8$ | $8.3\times10^8$ | | Wang et al. (2017) | Q | 80, 239 |
| WKRTWLVIFYOGIB-UHFFFAOYSA-N | $1.9\times10^{-1}$ | | Wang et al. (2017) | Q | 80, 240 |
| MCM:C5MEJPAN | $7.6\times10^7$ | | Wang et al. (2017) | Q | 80, 238 |
| $C_9H_{11}NO_8$ | $1.2\times10^9$ | | Wang et al. (2017) | Q | 80, 239 |
| KTOTUVLJTIGMKC-UHFFFAOYSA-N | $5.3\times10^{-1}$ | | Wang et al. (2017) | Q | 80, 240 |
| MCM:C8CO2OHPAN | $1.0\times10^8$ | | Wang et al. (2017) | Q | 80, 238 |
| $C_9H_{11}NO_8$ | $6.6\times10^8$ | | Wang et al. (2017) | Q | 80, 239 |
| UEMVFSSSLHUULZ-UHFFFAOYSA-N | $2.0\times10^{-1}$ | | Wang et al. (2017) | Q | 80, 240 |
| MCM:C8OHCO2PAN | $9.1\times10^7$ | | Wang et al. (2017) | Q | 80, 238 |
| $C_9H_{11}NO_8$ | $5.8\times10^8$ | | Wang et al. (2017) | Q | 80, 239 |
| GUTUSTBTBKWXHL-UHFFFAOYSA-N | $1.3\times10^{-1}$ | | Wang et al. (2017) | Q | 80, 240 |
| MCM:LIMALNO3 | $1.3\times10^8$ | 23000 | Wieser et al. (2023) | Q | 437 |
| $C_{10}H_{17}NO_6$ | $5.3\times10^6$ | | Wang et al. (2017) | Q | 80, 238 |
| FDBNDHKWTKEDSC-UHFFFAOYSA-N | $3.2\times10^7$ | | Wang et al. (2017) | Q | 80, 239 |
| | $3.1\times10^4$ | | Wang et al. (2017) | Q | 80, 240 |
| MCM:NLIMALOH | $5.3\times10^6$ | | Wang et al. (2017) | Q | 80, 238 |
| $C_{10}H_{17}NO_6$ | $4.8\times10^7$ | | Wang et al. (2017) | Q | 80, 239 |
| NNOYMJCUDKQQLW-UHFFFAOYSA-N | $2.7\times10^5$ | | Wang et al. (2017) | Q | 80, 240 |
| MCM:C117NO3 | $7.4\times10^6$ | | Wang et al. (2017) | Q | 80, 238 |
| $C_{11}H_{17}NO_6$ | $1.0\times10^7$ | | Wang et al. (2017) | Q | 80, 239 |
| VRGCJRKSASUXOA-UHFFFAOYSA-N | $2.8\times10^3$ | | Wang et al. (2017) | Q | 80, 240 |





Table A4.6: Nitrates ($RONO_2$) (...continued)

| Substance<br>Formula<br>(Trivial Name)<br>[CAS Registry Number]<br>InChIKey | $H_s^{cp}$<br>(at $T^\ominus$)<br>$\left[\dfrac{mol}{m^3\,Pa}\right]$ | $\dfrac{d\ln H_s^{cp}}{d(1/T)}$<br><br>[K] | Reference | Type | Note |
|---|---|---|---|---|---|
| MCM:C118NO3<br>$C_{11}H_{17}NO_7$<br>RHEKUCZWBZRVBN-UHFFFAOYSA-N | $2.2\times10^9$<br>$6.5\times10^8$<br>$3.2\times10^3$ | | Wang et al. (2017)<br>Wang et al. (2017)<br>Wang et al. (2017) | Q<br>Q<br>Q | 80, 238<br>80, 239<br>80, 240 |
| MCM:C1215NO3<br>$C_{12}H_{19}NO_7$<br>HIFZNEWSKKHKQF-UHFFFAOYSA-N | $1.8\times10^9$<br>$1.1\times10^{10}$<br>$2.6\times10^5$ | | Wang et al. (2017)<br>Wang et al. (2017)<br>Wang et al. (2017) | Q<br>Q<br>Q | 80, 238<br>80, 239<br>80, 240 |
| MCM:C128NO3<br>$C_{12}H_{19}NO_6$<br>XVMHBNQZULJJQH-UHFFFAOYSA-N | $6.5\times10^6$<br>$5.3\times10^7$<br>$1.1\times10^4$ | | Wang et al. (2017)<br>Wang et al. (2017)<br>Wang et al. (2017) | Q<br>Q<br>Q | 80, 238<br>80, 239<br>80, 240 |
| MCM:C1312NO3<br>$C_{13}H_{21}NO_6$<br>LCWPKTPKZZROTF-UHFFFAOYSA-N | $5.0\times10^6$<br>$1.4\times10^8$<br>$3.2\times10^4$ | | Wang et al. (2017)<br>Wang et al. (2017)<br>Wang et al. (2017) | Q<br>Q<br>Q | 80, 238<br>80, 239<br>80, 240 |
| MCM:BCALNO3<br>$C_{15}H_{25}NO_6$<br>KJHBLMRUHUAACD-UHFFFAOYSA-N | $4.9\times10^6$<br>$5.3\times10^7$<br>$8.7\times10^5$ | | Wang et al. (2017)<br>Wang et al. (2017)<br>Wang et al. (2017) | Q<br>Q<br>Q | 80, 238<br>80, 239<br>80, 240 |
| MCM:NBCALOH<br>$C_{15}H_{25}NO_6$<br>YUOPYTDNMDNILJ-UHFFFAOYSA-N | $4.9\times10^6$<br>$6.2\times10^7$<br>$6.9\times10^4$ | | Wang et al. (2017)<br>Wang et al. (2017)<br>Wang et al. (2017) | Q<br>Q<br>Q | 80, 238<br>80, 239<br>80, 240 |
| MCM:NO3CH2CO2H<br>$C_2H_3NO_5$<br>VHOVOPOBBVJMEP-UHFFFAOYSA-N | $1.6\times10^3$<br>$8.3\times10^4$<br>$1.7\times10^2$ | | Wang et al. (2017)<br>Wang et al. (2017)<br>Wang et al. (2017) | Q<br>Q<br>Q | 80, 238<br>80, 239<br>80, 240 |
| MCM:PRNO3CO2H<br>$C_3H_5NO_5$<br>DVOPCQDTEPKYKW-UHFFFAOYSA-N | $1.5\times10^3$<br>$9.1\times10^3$<br>$7.3\times10^1$ | | Wang et al. (2017)<br>Wang et al. (2017)<br>Wang et al. (2017) | Q<br>Q<br>Q | 80, 238<br>80, 239<br>80, 240 |
| MCM:MPRNO3CO2H<br>$C_4H_7NO_5$<br>MSBKTJKUKRRTOZ-UHFFFAOYSA-N | $8.3\times10^2$<br>$3.3\times10^3$<br>$3.2\times10^1$ | | Wang et al. (2017)<br>Wang et al. (2017)<br>Wang et al. (2017) | Q<br>Q<br>Q | 80, 238<br>80, 239<br>80, 240 |
| MCM:NC3CO2H<br>$C_4H_5NO_5$<br>IYBHCPMQXJSPPT-UHFFFAOYSA-N | $4.8\times10^3$<br>$1.4\times10^4$<br>$4.0\times10^2$ | | Wang et al. (2017)<br>Wang et al. (2017)<br>Wang et al. (2017) | Q<br>Q<br>Q | 80, 238<br>80, 239<br>80, 240 |
| MCM:C3MNO3CO2H<br>$C_5H_9NO_5$<br>CCKPHGGNTPAJHG-UHFFFAOYSA-N | $1.1\times10^3$<br>$2.2\times10^3$<br>$2.9\times10^1$ | | Wang et al. (2017)<br>Wang et al. (2017)<br>Wang et al. (2017) | Q<br>Q<br>Q | 80, 238<br>80, 239<br>80, 240 |
| MCM:C43NO3CO2H<br>$C_5H_9NO_5$<br>PAJLSJNKCRUQAA-UHFFFAOYSA-N | $6.8\times10^2$<br>$1.4\times10^3$<br>$2.4\times10^1$ | | Wang et al. (2017)<br>Wang et al. (2017)<br>Wang et al. (2017) | Q<br>Q<br>Q | 80, 238<br>80, 239<br>80, 240 |
| MCM:C4NO3CO2H<br>$C_5H_9NO_5$<br>LHOHBJOUNVJDSQ-UHFFFAOYSA-N | $9.8\times10^2$<br>$2.0\times10^3$<br>$2.7\times10^1$ | | Wang et al. (2017)<br>Wang et al. (2017)<br>Wang et al. (2017) | Q<br>Q<br>Q | 80, 238<br>80, 239<br>80, 240 |





Table A4.6: Nitrates ($RONO_2$) (... continued)

| Substance Formula (Trivial Name) [CAS Registry Number] InChIKey | $H_s^{cp}$ (at $T^\ominus$) $\left[\dfrac{\mathrm{mol}}{\mathrm{m^3\,Pa}}\right]$ | $\dfrac{\mathrm{d\ln} H_s^{cp}}{\mathrm{d}(1/T)}$ [K] | Reference | Type | Note |
|---|---|---|---|---|---|
| MCM:NC4CO2H | $3.2\times10^3$ | | Wang et al. (2017) | Q | 80, 238 |
| $C_5H_7NO_5$ | $9.6\times10^3$ | | Wang et al. (2017) | Q | 80, 239 |
| ZKTRJZJYWMAEEY-UHFFFAOYSA-N | $7.3\times10^1$ | | Wang et al. (2017) | Q | 80, 240 |
| MCM:C65NO3CO2H | $8.7\times10^2$ | | Wang et al. (2017) | Q | 80, 238 |
| $C_6H_{11}NO_5$ | $1.1\times10^3$ | | Wang et al. (2017) | Q | 80, 239 |
| JEDGAPUZBUOYQF-UHFFFAOYSA-N | $2.5\times10^1$ | | Wang et al. (2017) | Q | 80, 240 |
| MCM:C721PAN | $5.6\times10^5$ | | Wang et al. (2017) | Q | 80, 238 |
| $C_8H_{11}NO_7$ | $4.3\times10^6$ | | Wang et al. (2017) | Q | 80, 239 |
| OMUZIQDFMVGNTC-UHFFFAOYSA-N | $6.3\times10^1$ | | Wang et al. (2017) | Q | 80, 240 |
| MCM:C811NO3 | $1.5\times10^3$ | | Wang et al. (2017) | Q | 80, 238 |
| $C_8H_{13}NO_5$ | $1.7\times10^4$ | | Wang et al. (2017) | Q | 80, 239 |
| WYXUZPCNBFFDRO-UHFFFAOYSA-N | $5.6\times10^2$ | | Wang et al. (2017) | Q | 80, 240 |
| MCM:C823NO3 | $1.3\times10^3$ | | Wang et al. (2017) | Q | 80, 238 |
| $C_8H_{13}NO_5$ | $2.2\times10^4$ | | Wang et al. (2017) | Q | 80, 239 |
| QAJOXMTYFQHFPC-UHFFFAOYSA-N | $4.8\times10^2$ | | Wang et al. (2017) | Q | 80, 240 |
| MCM:C811PAN | $4.7\times10^5$ | | Wang et al. (2017) | Q | 80, 238 |
| $C_9H_{13}NO_7$ | $3.1\times10^6$ | | Wang et al. (2017) | Q | 80, 239 |
| BOYXRVFXWMBNIT-UHFFFAOYSA-N | $3.5\times10^2$ | | Wang et al. (2017) | Q | 80, 240 |
| MCM:C823PAN | $4.2\times10^5$ | | Wang et al. (2017) | Q | 80, 238 |
| $C_9H_{13}NO_7$ | $1.3\times10^6$ | | Wang et al. (2017) | Q | 80, 239 |
| GRHNTSHNCZZYLE-UHFFFAOYSA-N | $2.6\times10^2$ | | Wang et al. (2017) | Q | 80, 240 |
| MCM:C137NO3 | $1.3\times10^3$ | | Wang et al. (2017) | Q | 80, 238 |
| $C_{13}H_{21}NO_5$ | $3.8\times10^4$ | | Wang et al. (2017) | Q | 80, 239 |
| CQGOUXQAADNZJX-UHFFFAOYSA-N | $2.5\times10^3$ | | Wang et al. (2017) | Q | 80, 240 |
| MCM:C137PAN | $4.6\times10^5$ | | Wang et al. (2017) | Q | 80, 238 |
| $C_{14}H_{21}NO_7$ | $2.5\times10^6$ | | Wang et al. (2017) | Q | 80, 239 |
| KJDMPVAHMJBAAO-UHFFFAOYSA-N | $9.1\times10^2$ | | Wang et al. (2017) | Q | 80, 240 |
| MCM:MACRNBCO2H | $1.5\times10^5$ | | Wang et al. (2017) | Q | 80, 238 |
| $C_4H_7NO_6$ | $2.6\times10^6$ | | Wang et al. (2017) | Q | 80, 239 |
| WQOIEXBGIWRREH-UHFFFAOYSA-N | $3.6\times10^3$ | | Wang et al. (2017) | Q | 80, 240 |
| MCM:MACRNCO2H | $3.0\times10^6$ | | Wang et al. (2017) | Q | 80, 238 |
| $C_4H_7NO_6$ | $8.5\times10^5$ | | Wang et al. (2017) | Q | 80, 239 |
| GPIMHHNLFCYMAO-UHFFFAOYSA-N | $5.1\times10^3$ | | Wang et al. (2017) | Q | 80, 240 |
| MCM:C57NO3CO2H | $2.8\times10^9$ | | Wang et al. (2017) | Q | 80, 238 |
| $C_5H_9NO_7$ | $1.6\times10^8$ | | Wang et al. (2017) | Q | 80, 239 |
| VNPCVOYEGJTTSH-UHFFFAOYSA-N | $2.3\times10^5$ | | Wang et al. (2017) | Q | 80, 240 |
| MCM:C58NO3CO2H | $4.1\times10^8$ | | Wang et al. (2017) | Q | 80, 238 |
| $C_5H_9NO_7$ | $5.0\times10^8$ | | Wang et al. (2017) | Q | 80, 239 |
| QOEJXJLKIMUJIJ-UHFFFAOYSA-N | $1.0\times10^4$ | | Wang et al. (2017) | Q | 80, 240 |





Table A4.6: Nitrates ($RONO_2$) (...continued)

| Substance<br>Formula<br>(Trivial Name)<br>[CAS Registry Number]<br>InChIKey | $H_s^{cp}$<br>(at $T^{\ominus}$)<br>$\left[\dfrac{\text{mol}}{\text{m}^3\,\text{Pa}}\right]$ | $\dfrac{\mathrm{d}\ln H_s^{cp}}{\mathrm{d}(1/T)}$<br><br>[K] | Reference | Type | Note |
|---|---|---|---|---|---|
| MCM:INAHCO2H | $1.4\times10^8$ | | Wang et al. (2017) | Q | 80, 238 |
| $C_5H_9NO_7$ | $1.7\times10^9$ | | Wang et al. (2017) | Q | 80, 239 |
| KDGUUBLLRPAORR-UHFFFAOYSA-N | $5.0\times10^4$ | | Wang et al. (2017) | Q | 80, 240 |
| MCM:INAHPCO2H | $2.3\times10^{11}$ | | Wang et al. (2017) | Q | 80, 238 |
| $C_5H_9NO_8$ | $2.5\times10^9$ | | Wang et al. (2017) | Q | 80, 239 |
| CHIYBIMSWIBBRD-UHFFFAOYSA-N | $1.6\times10^5$ | | Wang et al. (2017) | Q | 80, 240 |
| MCM:INANCO2H | $3.6\times10^8$ | | Wang et al. (2017) | Q | 80, 238 |
| $C_5H_8N_2O_9$ | $3.6\times10^7$ | | Wang et al. (2017) | Q | 80, 239 |
| QEPBKMYMBRWSSY-UHFFFAOYSA-N | $4.5\times10^4$ | | Wang et al. (2017) | Q | 80, 240 |
| MCM:INB1HPCO2H | $2.0\times10^{11}$ | | Wang et al. (2017) | Q | 80, 238 |
| $C_5H_9NO_8$ | $2.9\times10^8$ | | Wang et al. (2017) | Q | 80, 239 |
| XWZUZABUMCCUCE-UHFFFAOYSA-N | $1.8\times10^5$ | | Wang et al. (2017) | Q | 80, 240 |
| MCM:INB1NACO2H | $3.0\times10^8$ | | Wang et al. (2017) | Q | 80, 238 |
| $C_5H_8N_2O_9$ | $5.1\times10^6$ | | Wang et al. (2017) | Q | 80, 239 |
| FONBWUWRCYYCPB-UHFFFAOYSA-N | $1.8\times10^3$ | | Wang et al. (2017) | Q | 80, 240 |
| MCM:INB1NBCO2H | $3.0\times10^8$ | | Wang et al. (2017) | Q | 80, 238 |
| $C_5H_8N_2O_9$ | $7.3\times10^6$ | | Wang et al. (2017) | Q | 80, 239 |
| YGORAHXKOXOIMV-UHFFFAOYSA-N | $2.9\times10^4$ | | Wang et al. (2017) | Q | 80, 240 |
| MCM:INCNCO2H | $3.6\times10^8$ | | Wang et al. (2017) | Q | 80, 238 |
| $C_5H_8N_2O_9$ | $2.0\times10^7$ | | Wang et al. (2017) | Q | 80, 239 |
| LXTUVHDUSAOJOD-UHFFFAOYSA-N | $1.6\times10^3$ | | Wang et al. (2017) | Q | 80, 240 |
| MCM:C139NO3 | $6.6\times10^7$ | | Wang et al. (2017) | Q | 80, 238 |
| $C_{13}H_{21}NO_7$ | $1.9\times10^8$ | | Wang et al. (2017) | Q | 80, 239 |
| UFNZJLDHNJULFY-UHFFFAOYSA-N | $6.5\times10^4$ | | Wang et al. (2017) | Q | 80, 240 |
| MCM:CONM2CO2H | $7.4\times10^5$ | | Wang et al. (2017) | Q | 80, 238 |
| $C_4H_5NO_6$ | $8.3\times10^4$ | | Wang et al. (2017) | Q | 80, 239 |
| LTGNKSWVEYSUMY-UHFFFAOYSA-N | $3.8\times10^1$ | | Wang et al. (2017) | Q | 80, 240 |
| MCM:MMALNACO2H | $1.2\times10^8$ | | Wang et al. (2017) | Q | 80, 238 |
| $C_5H_7NO_7$ | $4.5\times10^7$ | | Wang et al. (2017) | Q | 80, 239 |
| REHIBMNEJUPYOU-UHFFFAOYSA-N | $6.5\times10^2$ | | Wang et al. (2017) | Q | 80, 240 |
| MCM:MMALNBCO2H | $1.2\times10^8$ | | Wang et al. (2017) | Q | 80, 238 |
| $C_5H_7NO_7$ | $1.9\times10^7$ | | Wang et al. (2017) | Q | 80, 239 |
| RAMGTVAYJVLLRG-UHFFFAOYSA-N | $5.9\times10^3$ | | Wang et al. (2017) | Q | 80, 240 |
| MCM:INANCOCO2H | $5.8\times10^7$ | | Wang et al. (2017) | Q | 80, 238 |
| $C_5H_6N_2O_9$ | $8.5\times10^6$ | | Wang et al. (2017) | Q | 80, 239 |
| XEVMEFHPLZUFAK-UHFFFAOYSA-N | $6.0\times10^1$ | | Wang et al. (2017) | Q | 80, 240 |
| MCM:C732NO3 | $4.7\times10^5$ | | Wang et al. (2017) | Q | 80, 238 |
| $C_7H_{11}NO_6$ | $3.7\times10^7$ | | Wang et al. (2017) | Q | 80, 239 |
| SXYZBZXZNRYERD-UHFFFAOYSA-N | $4.9\times10^4$ | | Wang et al. (2017) | Q | 80, 240 |





Table A4.6: Nitrates ($RONO_2$) (...continued)

| Substance Formula (Trivial Name) [CAS Registry Number] InChIKey | $H_s^{cp}$ (at $T^{\ominus}$) $\left[\dfrac{\mathrm{mol}}{\mathrm{m^3\,Pa}}\right]$ | $\dfrac{\mathrm{d}\ln H_s^{cp}}{\mathrm{d}(1/T)}$ [K] | Reference | Type | Note |
|---|---|---|---|---|---|
| MCM:C732PAN | $1.7\times10^8$ | | Wang et al. (2017) | Q | 80, 238 |
| $C_8H_{11}NO_8$ | $1.7\times10^9$ | | Wang et al. (2017) | Q | 80, 239 |
| WNUKRJOERGBBRT-UHFFFAOYSA-N | $2.0\times10^4$ | | Wang et al. (2017) | Q | 80, 240 |
| MCM:C1211NO3 | $5.1\times10^5$ | | Wang et al. (2017) | Q | 80, 238 |
| $C_{12}H_{19}NO_6$ | $5.6\times10^7$ | | Wang et al. (2017) | Q | 80, 239 |
| DQVGSELEPUBBSS-UHFFFAOYSA-N | $2.9\times10^5$ | | Wang et al. (2017) | Q | 80, 240 |
| MCM:C1211PAN | $1.6\times10^8$ | | Wang et al. (2017) | Q | 80, 238 |
| $C_{13}H_{19}NO_8$ | $4.2\times10^9$ | | Wang et al. (2017) | Q | 80, 239 |
| IROGWMVYBREROI-UHFFFAOYSA-N | $9.8\times10^3$ | | Wang et al. (2017) | Q | 80, 240 |
| MCM:C813NO3 | $8.5\times10^8$ | | Wang et al. (2017) | Q | 80, 238 |
| $C_8H_{13}NO_7$ | $4.6\times10^7$ | | Wang et al. (2017) | Q | 80, 239 |
| GYZVFQOTJWTWBE-UHFFFAOYSA-N | $7.1\times10^5$ | | Wang et al. (2017) | Q | 80, 240 |
| MCM:C1212NO3 | $2.3\times10^{10}$ | | Wang et al. (2017) | Q | 80, 238 |
| $C_{12}H_{19}NO_8$ | $1.3\times10^{10}$ | | Wang et al. (2017) | Q | 80, 239 |
| NTNWJSADTNTFHH-UHFFFAOYSA-N | $3.0\times10^6$ | | Wang et al. (2017) | Q | 80, 240 |
| MCM:C1213NO3 | $2.6\times10^{11}$ | | Wang et al. (2017) | Q | 80, 238 |
| $C_{12}H_{19}NO_8$ | $1.8\times10^{11}$ | | Wang et al. (2017) | Q | 80, 239 |
| FRAJAIDKFGOMCR-UHFFFAOYSA-N | $3.6\times10^5$ | | Wang et al. (2017) | Q | 80, 240 |
| MCM:C1310NO3 | $7.3\times10^8$ | | Wang et al. (2017) | Q | 80, 238 |
| $C_{13}H_{21}NO_7$ | $1.4\times10^{10}$ | | Wang et al. (2017) | Q | 80, 239 |
| YNRLOWIUKIUEHR-UHFFFAOYSA-N | $3.0\times10^7$ | | Wang et al. (2017) | Q | 80, 240 |
| MCM:C151NO3 | $7.4\times10^8$ | | Wang et al. (2017) | Q | 80, 238 |
| $C_{15}H_{25}NO_7$ | $2.6\times10^{10}$ | | Wang et al. (2017) | Q | 80, 239 |
| BHPYCCPMXIBJRO-UHFFFAOYSA-N | $8.7\times10^8$ | | Wang et al. (2017) | Q | 80, 240 |
| MCM:CHOOCH2NO3 | $5.8$ | | Wang et al. (2017) | Q | 80, 238 |
| $C_2H_3NO_5$ | $1.4\times10^1$ | | Wang et al. (2017) | Q | 80, 239 |
| OAQKKIQKQLJUJT-UHFFFAOYSA-N | $2.3\times10^{-2}$ | | Wang et al. (2017) | Q | 80, 240 |
| MCM:CHOOMPAN | $1.8\times10^3$ | | Wang et al. (2017) | Q | 80, 238 |
| $C_3H_3NO_7$ | $1.6\times10^3$ | | Wang et al. (2017) | Q | 80, 239 |
| XZTQWYQUABOKPG-UHFFFAOYSA-N | $1.9\times10^{-2}$ | | Wang et al. (2017) | Q | 80, 240 |
| MCM:ETHFORMNO3 | $5.4$ | | Wang et al. (2017) | Q | 80, 238 |
| $C_3H_5NO_5$ | $3.5$ | | Wang et al. (2017) | Q | 80, 239 |
| DOXRBPLNMOIQGP-UHFFFAOYSA-N | $1.6\times10^{-2}$ | | Wang et al. (2017) | Q | 80, 240 |
| MCM:METACETNO3 | $3.9$ | | Wang et al. (2017) | Q | 80, 238 |
| $C_3H_5NO_5$ | $1.9\times10^1$ | | Wang et al. (2017) | Q | 80, 239 |
| HRSAJZJMUADCNM-UHFFFAOYSA-N | $3.6\times10^{-2}$ | | Wang et al. (2017) | Q | 80, 240 |
| MCM:MMCFNO3 | $1.8\times10^3$ | | Wang et al. (2017) | Q | 80, 238 |
| $C_3H_3NO_7$ | $1.1\times10^4$ | | Wang et al. (2017) | Q | 80, 239 |
| IFMSYUVMGRDEMV-UHFFFAOYSA-N | $7.3\times10^{-2}$ | | Wang et al. (2017) | Q | 80, 240 |



Table A4.6: Nitrates ($RONO_2$) (. . . continued)

| Substance<br>Formula<br>(Trivial Name)<br>[CAS Registry Number]<br>InChIKey | $H_s^{cp}$<br>(at $T^\ominus$)<br>$\left[\dfrac{\mathrm{mol}}{\mathrm{m^3\,Pa}}\right]$ | $\dfrac{\mathrm{d\ln} H_s^{cp}}{\mathrm{d}(1/T)}$<br><br>[K] | Reference | Type | Note |
|---|---|---|---|---|---|
| MCM:ACEC2H4NO3 | 3.2 | | Wang et al. (2017) | Q | 80, 238 |
| $C_4H_7NO_5$ | $2.5\times10^1$ | | Wang et al. (2017) | Q | 80, 239 |
| QOXSFWPQUXLCPI-UHFFFAOYSA-N | $9.8\times10^{-1}$ | | Wang et al. (2017) | Q | 80, 240 |
| MCM:ACETMEPAN | $1.2\times10^3$ | | Wang et al. (2017) | Q | 80, 238 |
| $C_4H_5NO_7$ | $1.8\times10^3$ | | Wang et al. (2017) | Q | 80, 239 |
| QMSGSFOWAUFKPL-UHFFFAOYSA-N | $1.7\times10^{-2}$ | | Wang et al. (2017) | Q | 80, 240 |
| MCM:COO2C3PAN | $1.2\times10^3$ | | Wang et al. (2017) | Q | 80, 238 |
| $C_4H_5NO_7$ | $1.6\times10^3$ | | Wang et al. (2017) | Q | 80, 239 |
| YQMLKDOXYNVTNM-UHFFFAOYSA-N | $7.1\times10^{-2}$ | | Wang et al. (2017) | Q | 80, 240 |
| MCM:ETACETNO3 | 3.6 | | Wang et al. (2017) | Q | 80, 238 |
| $C_4H_7NO_5$ | 3.5 | | Wang et al. (2017) | Q | 80, 239 |
| KORDSAOMZNGUAL-UHFFFAOYSA-N | $2.4\times10^{-2}$ | | Wang et al. (2017) | Q | 80, 240 |
| MCM:IPRFORMNO3 | 3.0 | | Wang et al. (2017) | Q | 80, 238 |
| $C_4H_7NO_5$ | $6.9\times10^{-1}$ | | Wang et al. (2017) | Q | 80, 239 |
| DZSICYNSKFSRMV-UHFFFAOYSA-N | $1.8\times10^{-2}$ | | Wang et al. (2017) | Q | 80, 240 |
| MCM:NBZFUOOH | $2.2\times10^8$ | | Wang et al. (2017) | Q | 80, 238 |
| $C_4H_5NO_7$ | $3.0\times10^6$ | | Wang et al. (2017) | Q | 80, 239 |
| TXWHLSUKLWEERR-UHFFFAOYSA-N | $2.5\times10^3$ | | Wang et al. (2017) | Q | 80, 240 |
| MCM:ACCOMEPAN | $5.3\times10^4$ | | Wang et al. (2017) | Q | 80, 238 |
| $C_5H_5NO_8$ | $5.1\times10^5$ | | Wang et al. (2017) | Q | 80, 239 |
| ZQOVYYMCAFZVPL-UHFFFAOYSA-N | 1.9 | | Wang et al. (2017) | Q | 80, 240 |
| MCM:ACETC2PAN | $1.0\times10^3$ | | Wang et al. (2017) | Q | 80, 238 |
| $C_5H_7NO_7$ | $2.1\times10^3$ | | Wang et al. (2017) | Q | 80, 239 |
| VMVPPLVFXDAZML-UHFFFAOYSA-N | $6.5\times10^{-1}$ | | Wang et al. (2017) | Q | 80, 240 |
| MCM:COO2C4PAN | $1.0\times10^3$ | | Wang et al. (2017) | Q | 80, 238 |
| $C_5H_7NO_7$ | $1.7\times10^3$ | | Wang et al. (2017) | Q | 80, 239 |
| GBAXNZCSCJEGAE-UHFFFAOYSA-N | $1.6\times10^{-1}$ | | Wang et al. (2017) | Q | 80, 240 |
| MCM:IPRACBNO3 | 2.8 | | Wang et al. (2017) | Q | 80, 238 |
| $C_5H_9NO_5$ | 8.1 | | Wang et al. (2017) | Q | 80, 239 |
| ZEVCPCUWRCOHCL-UHFFFAOYSA-N | $3.4\times10^{-1}$ | | Wang et al. (2017) | Q | 80, 240 |
| MCM:IPRACBPAN | $1.1\times10^3$ | | Wang et al. (2017) | Q | 80, 238 |
| $C_5H_7NO_7$ | $3.9\times10^2$ | | Wang et al. (2017) | Q | 80, 239 |
| UJCZQTHKWBYHPW-UHFFFAOYSA-N | $2.2\times10^{-2}$ | | Wang et al. (2017) | Q | 80, 240 |
| MCM:IPRACNO3 | 2.0 | | Wang et al. (2017) | Q | 80, 238 |
| $C_5H_9NO_5$ | $8.0\times10^{-1}$ | | Wang et al. (2017) | Q | 80, 239 |
| JAQKAPGOMZJVBI-UHFFFAOYSA-N | $2.2\times10^{-2}$ | | Wang et al. (2017) | Q | 80, 240 |
| MCM:MTBEAALNO3 | 2.7 | | Wang et al. (2017) | Q | 80, 238 |
| $C_5H_9NO_5$ | 2.5 | | Wang et al. (2017) | Q | 80, 239 |
| NIMOOIUOXYQIIM-UHFFFAOYSA-N | $1.4\times10^{-1}$ | | Wang et al. (2017) | Q | 80, 240 |





Table A4.6: Nitrates ($RONO_2$) (...continued)

| Substance<br>Formula<br>(Trivial Name)<br>[CAS Registry Number]<br>InChIKey | $H_s^{cp}$<br>(at $T^\ominus$)<br>$\left[\dfrac{\text{mol}}{\text{m}^3\,\text{Pa}}\right]$ | $\dfrac{\mathrm{d}\ln H_s^{cp}}{\mathrm{d}(1/T)}$<br><br>[K] | Reference | Type | Note |
|---|---|---|---|---|---|
| MCM:MTBEAALPAN<br>$C_5H_7NO_7$<br>WKTKTTPKCZOXAQ-UHFFFAOYSA-N | $9.6\times10^2$<br>$6.5\times10^1$<br>$1.1\times10^{-2}$ | | Wang et al. (2017)<br>Wang et al. (2017)<br>Wang et al. (2017) | Q<br>Q<br>Q | 80, 238<br>80, 239<br>80, 240 |
| MCM:NPRACANO3<br>$C_5H_9NO_5$<br>FPQZKAXTSOKJRB-UHFFFAOYSA-N | 2.8<br>8.1<br>$3.4\times10^{-1}$ | | Wang et al. (2017)<br>Wang et al. (2017)<br>Wang et al. (2017) | Q<br>Q<br>Q | 80, 238<br>80, 239<br>80, 240 |
| MCM:NPRACBNO3<br>$C_5H_9NO_5$<br>HTKMYQNNJMTEJQ-UHFFFAOYSA-N | 2.8<br>1.9<br>$1.6\times10^{-2}$ | | Wang et al. (2017)<br>Wang et al. (2017)<br>Wang et al. (2017) | Q<br>Q<br>Q | 80, 238<br>80, 239<br>80, 240 |
| MCM:NPRACCNO3<br>$C_5H_9NO_5$<br>PSJLJILPBUYXRJ-UHFFFAOYSA-N | 2.8<br>$1.5\times10^1$<br>3.1 | | Wang et al. (2017)<br>Wang et al. (2017)<br>Wang et al. (2017) | Q<br>Q<br>Q | 80, 238<br>80, 239<br>80, 240 |
| MCM:NPXYFUOOH<br>$C_5H_7NO_7$<br>ARRSNVVNYKGUFK-UHFFFAOYSA-N | $1.2\times10^8$<br>$8.9\times10^5$<br>$5.6\times10^2$ | | Wang et al. (2017)<br>Wang et al. (2017)<br>Wang et al. (2017) | Q<br>Q<br>Q | 80, 238<br>80, 239<br>80, 240 |
| MCM:NTLFUOOH<br>$C_5H_7NO_7$<br>OAXQEILKHMZHNV-UHFFFAOYSA-N | $2.0\times10^8$<br>$2.2\times10^6$<br>$1.1\times10^3$ | | Wang et al. (2017)<br>Wang et al. (2017)<br>Wang et al. (2017) | Q<br>Q<br>Q | 80, 238<br>80, 239<br>80, 240 |
| MCM:MCOOTBNO3<br>$C_6H_{11}NO_5$<br>BCWPDUXSEGSVPT-UHFFFAOYSA-N | 1.6<br>2.9<br>$1.2\times10^{-1}$ | | Wang et al. (2017)<br>Wang et al. (2017)<br>Wang et al. (2017) | Q<br>Q<br>Q | 80, 238<br>80, 239<br>80, 240 |
| MCM:NBUACANO3<br>$C_6H_{11}NO_5$<br>GNPRUWODPALWHJ-UHFFFAOYSA-N | 2.6<br>6.9<br>1.4 | | Wang et al. (2017)<br>Wang et al. (2017)<br>Wang et al. (2017) | Q<br>Q<br>Q | 80, 238<br>80, 239<br>80, 240 |
| MCM:NBUACBNO3<br>$C_6H_{11}NO_5$<br>UFOXFUDLAIUTHC-UHFFFAOYSA-N | 2.6<br>5.0<br>$4.0\times10^{-1}$ | | Wang et al. (2017)<br>Wang et al. (2017)<br>Wang et al. (2017) | Q<br>Q<br>Q | 80, 238<br>80, 239<br>80, 240 |
| MCM:NBUACCNO3<br>$C_6H_{11}NO_5$<br>PYXJHVBEHHCABK-UHFFFAOYSA-N | 2.6<br>1.4<br>$1.3\times10^{-2}$ | | Wang et al. (2017)<br>Wang et al. (2017)<br>Wang et al. (2017) | Q<br>Q<br>Q | 80, 238<br>80, 239<br>80, 240 |
| MCM:NDMMALYOOH<br>$C_6H_7NO_8$<br>RZSLUQVSVWAZFJ-UHFFFAOYSA-N | $2.2\times10^{11}$<br>$9.8\times10^7$<br>$2.5\times10^2$ | | Wang et al. (2017)<br>Wang et al. (2017)<br>Wang et al. (2017) | Q<br>Q<br>Q | 80, 238<br>80, 239<br>80, 240 |
| MCM:NEBFUOOH<br>$C_6H_9NO_7$<br>CZSCMTWZZQSKMX-UHFFFAOYSA-N | $1.8\times10^8$<br>$1.4\times10^6$<br>$9.8\times10^2$ | | Wang et al. (2017)<br>Wang et al. (2017)<br>Wang et al. (2017) | Q<br>Q<br>Q | 80, 238<br>80, 239<br>80, 240 |
| MCM:NMXYFUOOH<br>$C_6H_9NO_7$<br>URYSYTGABCBDCM-UHFFFAOYSA-N | $1.1\times10^8$<br>$4.9\times10^5$<br>$6.3\times10^2$ | | Wang et al. (2017)<br>Wang et al. (2017)<br>Wang et al. (2017) | Q<br>Q<br>Q | 80, 238<br>80, 239<br>80, 240 |





Table A4.6: Nitrates ($RONO_2$) (... continued)

| Substance<br>Formula<br>(Trivial Name)<br>[CAS Registry Number]<br>InChIKey | $H_s^{cp}$<br>(at $T^\ominus$)<br>$\left[\dfrac{\mathrm{mol}}{\mathrm{m^3\,Pa}}\right]$ | $\dfrac{\mathrm{d}\ln H_s^{cp}}{\mathrm{d}(1/T)}$<br><br>[K] | Reference | Type | Note |
|---|---|---|---|---|---|
| MCM:NOXYFUOOH | $6.8\times10^7$ | | Wang et al. (2017) | Q | 80, 238 |
| $C_6H_9NO_7$ | $3.2\times10^5$ | | Wang et al. (2017) | Q | 80, 239 |
| CPPAIMSHWPDUBG-UHFFFAOYSA-N | $2.4\times10^3$ | | Wang et al. (2017) | Q | 80, 240 |
| MCM:NTMB1FUOOH | $1.1\times10^8$ | | Wang et al. (2017) | Q | 80, 238 |
| $C_6H_9NO_7$ | $1.7\times10^7$ | | Wang et al. (2017) | Q | 80, 239 |
| WRMOSOIHGVCRMO-UHFFFAOYSA-N | $2.3\times10^3$ | | Wang et al. (2017) | Q | 80, 240 |
| MCM:PRCOOMPAN | $8.9\times10^2$ | | Wang et al. (2017) | Q | 80, 238 |
| $C_6H_9NO_7$ | $5.6\times10^2$ | | Wang et al. (2017) | Q | 80, 239 |
| NSDDFRROKVQLRJ-UHFFFAOYSA-N | $1.4\times10^{-2}$ | | Wang et al. (2017) | Q | 80, 240 |
| MCM:SBUACANO3 | 1.6 | | Wang et al. (2017) | Q | 80, 238 |
| $C_6H_{11}NO_5$ | $5.3\times10^{-1}$ | | Wang et al. (2017) | Q | 80, 239 |
| FWAMAOJLVLFAOA-UHFFFAOYSA-N | $1.9\times10^{-2}$ | | Wang et al. (2017) | Q | 80, 240 |
| MCM:SBUACBNO3 | 3.0 | | Wang et al. (2017) | Q | 80, 238 |
| $C_6H_{11}NO_5$ | 4.0 | | Wang et al. (2017) | Q | 80, 239 |
| FVMFJLQQOIKLHP-UHFFFAOYSA-N | $2.8\times10^{-1}$ | | Wang et al. (2017) | Q | 80, 240 |
| MCM:TBUACPAN | $6.3\times10^2$ | | Wang et al. (2017) | Q | 80, 238 |
| $C_6H_9NO_7$ | $7.3\times10^1$ | | Wang et al. (2017) | Q | 80, 239 |
| WTCUHXTZKDYOTB-UHFFFAOYSA-N | $2.6\times10^{-2}$ | | Wang et al. (2017) | Q | 80, 240 |
| MCM:NIPBFUOOH | $1.7\times10^8$ | | Wang et al. (2017) | Q | 80, 238 |
| $C_7H_{11}NO_7$ | $1.3\times10^6$ | | Wang et al. (2017) | Q | 80, 239 |
| MJSJHBOZYWKETN-UHFFFAOYSA-N | $5.4\times10^2$ | | Wang et al. (2017) | Q | 80, 240 |
| MCM:NMEBFUOOH | $1.0\times10^8$ | | Wang et al. (2017) | Q | 80, 238 |
| $C_7H_{11}NO_7$ | $3.2\times10^5$ | | Wang et al. (2017) | Q | 80, 239 |
| LKMBYBKEMVMSAU-UHFFFAOYSA-N | $3.6\times10^2$ | | Wang et al. (2017) | Q | 80, 240 |
| MCM:NPBFUOOH | $1.5\times10^8$ | | Wang et al. (2017) | Q | 80, 238 |
| $C_7H_{11}NO_7$ | $1.1\times10^6$ | | Wang et al. (2017) | Q | 80, 239 |
| DGKCEZHRNXTHAZ-UHFFFAOYSA-N | $4.0\times10^2$ | | Wang et al. (2017) | Q | 80, 240 |
| MCM:NTMB2FUOOH | $6.3\times10^7$ | | Wang et al. (2017) | Q | 80, 238 |
| $C_7H_{11}NO_7$ | $1.8\times10^5$ | | Wang et al. (2017) | Q | 80, 239 |
| IRLXCLTZUJZGMJ-UHFFFAOYSA-N | $2.4\times10^2$ | | Wang et al. (2017) | Q | 80, 240 |
| MCM:C1013NO3 | 3.1 | | Wang et al. (2017) | Q | 80, 238 |
| $C_{10}H_{17}NO_5$ | 5.4 | | Wang et al. (2017) | Q | 80, 239 |
| TWKDJCLTDHUPBZ-UHFFFAOYSA-N | 3.2 | | Wang et al. (2017) | Q | 80, 240 |
| MCM:C1013PAN | $9.8\times10^2$ | | Wang et al. (2017) | Q | 80, 238 |
| $C_{11}H_{17}NO_7$ | $4.8\times10^2$ | | Wang et al. (2017) | Q | 80, 239 |
| QVQKWVAIHQGLJY-UHFFFAOYSA-N | 1.3 | | Wang et al. (2017) | Q | 80, 240 |
| MCM:C1014NO3 | $7.8\times10^2$ | | Wang et al. (2017) | Q | 80, 238 |
| $C_{10}H_{17}NO_6$ | $1.9\times10^3$ | | Wang et al. (2017) | Q | 80, 239 |
| OQZDTLQTCPZSBL-UHFFFAOYSA-N | $7.8\times10^1$ | | Wang et al. (2017) | Q | 80, 240 |



Table A4.6: Nitrates ($RONO_2$) (...continued)

| Substance Formula (Trivial Name) [CAS Registry Number] InChIKey | $H_s^{cp}$ (at $T^\ominus$) $\left[\dfrac{mol}{m^3\,Pa}\right]$ | $\dfrac{d\ln H_s^{cp}}{d(1/T)}$ [K] | Reference | Type | Note |
|---|---|---|---|---|---|
| MCM:C152NO3 | $3.0\times10^6$ | | Wang et al. (2017) | Q | 80, 238 |
| $C_{15}H_{25}NO_7$ | $1.3\times10^7$ | | Wang et al. (2017) | Q | 80, 239 |
| VUWXYQDBWQFZDF-UHFFFAOYSA-N | $7.1\times10^4$ | | Wang et al. (2017) | Q | 80, 240 |
| MCM:NBZFUONE | $1.9\times10^6$ | | Wang et al. (2017) | Q | 80, 238 |
| $C_4H_3NO_6$ | $4.3\times10^6$ | | Wang et al. (2017) | Q | 80, 239 |
| LXADUFLCRLLGHU-UHFFFAOYSA-N | $8.3\times10^1$ | | Wang et al. (2017) | Q | 80, 240 |
| MCM:C23O3CPAN | $7.6\times10^5$ | | Wang et al. (2017) | Q | 80, 238 |
| $C_5H_5NO_8$ | $2.8\times10^5$ | | Wang et al. (2017) | Q | 80, 239 |
| BCRDXKGRUUAZSU-UHFFFAOYSA-N | $3.0\times10^{-1}$ | | Wang et al. (2017) | Q | 80, 240 |
| MCM:ACBUONANO3 | $1.7\times10^3$ | | Wang et al. (2017) | Q | 80, 238 |
| $C_6H_9NO_6$ | $2.2\times10^3$ | | Wang et al. (2017) | Q | 80, 239 |
| PDSBHIWVTCGSAN-UHFFFAOYSA-N | $9.1$ | | Wang et al. (2017) | Q | 80, 240 |
| MCM:C23O3MCPAN | $7.1\times10^5$ | | Wang et al. (2017) | Q | 80, 238 |
| $C_6H_7NO_8$ | $6.0\times10^4$ | | Wang et al. (2017) | Q | 80, 239 |
| MJYWDTPJLIMPPV-UHFFFAOYSA-N | $2.3\times10^{-1}$ | | Wang et al. (2017) | Q | 80, 240 |
| MCM:NEBFUONE | $1.4\times10^6$ | | Wang et al. (2017) | Q | 80, 238 |
| $C_6H_7NO_6$ | $2.3\times10^6$ | | Wang et al. (2017) | Q | 80, 239 |
| LZGWKDALCOZNIJ-UHFFFAOYSA-N | $8.1$ | | Wang et al. (2017) | Q | 80, 240 |
| MCM:C23O3ECPAN | $5.5\times10^5$ | | Wang et al. (2017) | Q | 80, 238 |
| $C_7H_9NO_8$ | $3.2\times10^4$ | | Wang et al. (2017) | Q | 80, 239 |
| SOMJWLUSPZTMJM-UHFFFAOYSA-N | $1.5\times10^{-1}$ | | Wang et al. (2017) | Q | 80, 240 |
| MCM:NIPBFUONE | $1.3\times10^6$ | | Wang et al. (2017) | Q | 80, 238 |
| $C_7H_9NO_6$ | $2.0\times10^6$ | | Wang et al. (2017) | Q | 80, 239 |
| OEYHEYBMUYSQHE-UHFFFAOYSA-N | $4.8$ | | Wang et al. (2017) | Q | 80, 240 |
| MCM:NPBFUONE | $1.3\times10^6$ | | Wang et al. (2017) | Q | 80, 238 |
| $C_7H_9NO_6$ | $1.7\times10^6$ | | Wang et al. (2017) | Q | 80, 239 |
| WTNUMEKWPLLNDJ-UHFFFAOYSA-N | $6.6$ | | Wang et al. (2017) | Q | 80, 240 |
| MCM:CH3OCH2NO3 | $3.4\times10^{-1}$ | | Wang et al. (2017) | Q | 80, 238 |
| $C_2H_5NO_4$ | $3.4\times10^{-1}$ | | Wang et al. (2017) | Q | 80, 239 |
| VHAYDBAVTGOZLW-UHFFFAOYSA-N | $4.5\times10^{-3}$ | | Wang et al. (2017) | Q | 80, 240 |
| MCM:DMMANO3 | $8.1$ | | Wang et al. (2017) | Q | 80, 238 |
| $C_3H_7NO_5$ | $5.5$ | | Wang et al. (2017) | Q | 80, 239 |
| ZKGVKOFAVUIIEY-UHFFFAOYSA-N | $3.0\times10^{-2}$ | | Wang et al. (2017) | Q | 80, 240 |
| MCM:DMMBNO3 | $9.3$ | | Wang et al. (2017) | Q | 80, 238 |
| $C_3H_7NO_5$ | $1.0\times10^{-1}$ | | Wang et al. (2017) | Q | 80, 239 |
| ADJOFNYUECRONI-UHFFFAOYSA-N | $1.7\times10^{-2}$ | | Wang et al. (2017) | Q | 80, 240 |
| MCM:ETOMENO3 | $5.8\times10^{-1}$ | | Wang et al. (2017) | Q | 80, 238 |
| $C_3H_7NO_3$ | $6.6\times10^{-1}$ | | Wang et al. (2017) | Q | 80, 239 |
| ZMUPMMOTQUHRHQ-UHFFFAOYSA-N | $3.9\times10^{-1}$ | | Wang et al. (2017) | Q | 80, 240 |





Table A4.6: Nitrates ($RONO_2$) (...continued)

| Substance Formula (Trivial Name) [CAS Registry Number] InChIKey | $H_s^{cp}$ (at $T^{\ominus}$) $\left[\dfrac{\mathrm{mol}}{\mathrm{m^3\,Pa}}\right]$ | $\dfrac{\mathrm{d}\ln H_s^{cp}}{\mathrm{d}(1/T)}$ [K] | Reference | Type | Note |
|---|---|---|---|---|---|
| MCM:MEMOXYPAN | $1.1\times10^2$ | | Wang et al. (2017) | Q | 80, 238 |
| $C_3H_5NO_6$ | $2.2\times10^2$ | | Wang et al. (2017) | Q | 80, 239 |
| JBHGAGLULWBGCW-UHFFFAOYSA-N | $1.0\times10^{-2}$ | | Wang et al. (2017) | Q | 80, 240 |
| MCM:DIETETNO3 | $2.8\times10^{-1}$ | | Wang et al. (2017) | Q | 80, 238 |
| $C_4H_9NO_4$ | $4.2\times10^{-2}$ | | Wang et al. (2017) | Q | 80, 239 |
| CAMRTYJYAQGENN-UHFFFAOYSA-N | $7.3\times10^{-3}$ | | Wang et al. (2017) | Q | 80, 240 |
| MCM:ETOC2NO3 | $2.5\times10^{-1}$ | | Wang et al. (2017) | Q | 80, 238 |
| $C_4H_9NO_4$ | $2.5$ | | Wang et al. (2017) | Q | 80, 239 |
| GDNQXPDYGNUKII-UHFFFAOYSA-N | $4.3\times10^{-2}$ | | Wang et al. (2017) | Q | 80, 240 |
| MCM:ETOMEPAN | $9.3\times10^1$ | | Wang et al. (2017) | Q | 80, 238 |
| $C_4H_7NO_6$ | $1.1\times10^2$ | | Wang et al. (2017) | Q | 80, 239 |
| CCYZAMBLJVQHMC-UHFFFAOYSA-N | $1.3\times10^{-2}$ | | Wang et al. (2017) | Q | 80, 240 |
| MCM:IPROC21NO3 | $2.6\times10^{-1}$ | | Wang et al. (2017) | Q | 80, 238 |
| $C_5H_{11}NO_4$ | $2.0\times10^{-2}$ | | Wang et al. (2017) | Q | 80, 239 |
| YQGFTPNVRZNXAP-UHFFFAOYSA-N | $8.0\times10^{-3}$ | | Wang et al. (2017) | Q | 80, 240 |
| MCM:MTBEANO3 | $1.5\times10^{-1}$ | | Wang et al. (2017) | Q | 80, 238 |
| $C_5H_{11}NO_4$ | $2.7\times10^{-2}$ | | Wang et al. (2017) | Q | 80, 239 |
| NXHUJDLNRCHXIE-UHFFFAOYSA-N | $8.3\times10^{-3}$ | | Wang et al. (2017) | Q | 80, 240 |
| MCM:MTBEBNO3 | $1.5\times10^{-1}$ | | Wang et al. (2017) | Q | 80, 238 |
| $C_5H_{11}NO_4$ | $6.2\times10^{-1}$ | | Wang et al. (2017) | Q | 80, 239 |
| HWZCMFJGZXOLOM-UHFFFAOYSA-N | $4.1\times10^{-2}$ | | Wang et al. (2017) | Q | 80, 240 |
| MCM:MTBEBPAN | $5.4\times10^1$ | | Wang et al. (2017) | Q | 80, 238 |
| $C_5H_9NO_6$ | $1.4\times10^1$ | | Wang et al. (2017) | Q | 80, 239 |
| PQCDPGWYIKXNFH-UHFFFAOYSA-N | $5.5\times10^{-3}$ | | Wang et al. (2017) | Q | 80, 240 |
| MCM:BOXMPAN | $6.0\times10^1$ | | Wang et al. (2017) | Q | 80, 238 |
| $C_6H_{11}NO_6$ | $5.1\times10^1$ | | Wang et al. (2017) | Q | 80, 239 |
| CPESEOWHBGRFRT-UHFFFAOYSA-N | $3.2\times10^{-3}$ | | Wang et al. (2017) | Q | 80, 240 |
| MCM:DIIPRETNO3 | $1.4\times10^{-1}$ | | Wang et al. (2017) | Q | 80, 238 |
| $C_6H_{13}NO_4$ | $4.9\times10^{-3}$ | | Wang et al. (2017) | Q | 80, 239 |
| RYKOUPKPIPKSCM-UHFFFAOYSA-N | $1.5\times10^{-2}$ | | Wang et al. (2017) | Q | 80, 240 |
| MCM:ETBEANO3 | $1.2\times10^{-1}$ | | Wang et al. (2017) | Q | 80, 238 |
| $C_6H_{13}NO_4$ | $3.2\times10^{-1}$ | | Wang et al. (2017) | Q | 80, 239 |
| WGHLTNOFPVNDHK-UHFFFAOYSA-N | $4.1\times10^{-2}$ | | Wang et al. (2017) | Q | 80, 240 |
| MCM:ETBEAPAN | $4.8\times10^1$ | | Wang et al. (2017) | Q | 80, 238 |
| $C_6H_{11}NO_6$ | $9.1$ | | Wang et al. (2017) | Q | 80, 239 |
| ZFJPSAIQYMAFAB-UHFFFAOYSA-N | $3.6\times10^{-3}$ | | Wang et al. (2017) | Q | 80, 240 |
| MCM:ETBEBNO3 | $1.4\times10^{-1}$ | | Wang et al. (2017) | Q | 80, 238 |
| $C_6H_{13}NO_4$ | $7.1\times10^{-3}$ | | Wang et al. (2017) | Q | 80, 239 |
| BIDDBAQEKWTVIC-UHFFFAOYSA-N | $1.1\times10^{-2}$ | | Wang et al. (2017) | Q | 80, 240 |



Table A4.6: Nitrates ($RONO_2$) (...continued)

| Substance Formula (Trivial Name) [CAS Registry Number] InChIKey | $H_s^{cp}$ (at $T^{\ominus}$) $\left[\dfrac{\mathrm{mol}}{\mathrm{m}^3\,\mathrm{Pa}}\right]$ | $\dfrac{\mathrm{d}\ln H_s^{cp}}{\mathrm{d}(1/T)}$ [K] | Reference | Type | Note |
|---|---|---|---|---|---|
| MCM:ETBECNO3 | $1.2\times10^{-1}$ | | Wang et al. (2017) | Q | 80, 238 |
| $C_6H_{13}NO_4$ | $4.0\times10^{-1}$ | | Wang et al. (2017) | Q | 80, 239 |
| MGOGEALWXAMIEO-UHFFFAOYSA-N | $5.0\times10^{-2}$ | | Wang et al. (2017) | Q | 80, 240 |
| MCM:ETBECPAN | $4.8\times10^{1}$ | | Wang et al. (2017) | Q | 80, 238 |
| $C_6H_{11}NO_6$ | $2.3\times10^{1}$ | | Wang et al. (2017) | Q | 80, 239 |
| JICVHVAZXJWJCM-UHFFFAOYSA-N | $2.1\times10^{-2}$ | | Wang et al. (2017) | Q | 80, 240 |
| MCM:IPROMC2NO3 | $2.0\times10^{-1}$ | | Wang et al. (2017) | Q | 80, 238 |
| $C_6H_{13}NO_4$ | $4.6\times10^{-1}$ | | Wang et al. (2017) | Q | 80, 239 |
| XJQWDDGHWZGMDK-UHFFFAOYSA-N | $2.9\times10^{-2}$ | | Wang et al. (2017) | Q | 80, 240 |
| MCM:IPROMCPAN | $8.3\times10^{1}$ | | Wang et al. (2017) | Q | 80, 238 |
| $C_6H_{11}NO_6$ | $2.0\times10^{1}$ | | Wang et al. (2017) | Q | 80, 239 |
| SFYHCJJQTXJNNN-UHFFFAOYSA-N | $3.9\times10^{-3}$ | | Wang et al. (2017) | Q | 80, 240 |
| MCM:MO2EOLANO3 | $1.2\times10^{3}$ | | Wang et al. (2017) | Q | 80, 238 |
| $C_3H_7NO_5$ | $5.6\times10^{2}$ | | Wang et al. (2017) | Q | 80, 239 |
| FJVLHDUAJQRZPG-UHFFFAOYSA-N | $3.2$ | | Wang et al. (2017) | Q | 80, 240 |
| MCM:MO2EOLBNO3 | $1.0\times10^{3}$ | | Wang et al. (2017) | Q | 80, 238 |
| $C_3H_7NO_5$ | $1.3\times10^{3}$ | | Wang et al. (2017) | Q | 80, 239 |
| QPXHYMSZINTGRC-UHFFFAOYSA-N | $2.8\times10^{1}$ | | Wang et al. (2017) | Q | 80, 240 |
| MCM:EOX2OLANO3 | $1.0\times10^{3}$ | | Wang et al. (2017) | Q | 80, 238 |
| $C_4H_9NO_5$ | $3.6\times10^{2}$ | | Wang et al. (2017) | Q | 80, 239 |
| UVLBVBPSPHSYND-UHFFFAOYSA-N | $1.9$ | | Wang et al. (2017) | Q | 80, 240 |
| MCM:EOX2OLBNO3 | $9.6\times10^{2}$ | | Wang et al. (2017) | Q | 80, 238 |
| $C_4H_9NO_5$ | $3.2\times10^{2}$ | | Wang et al. (2017) | Q | 80, 239 |
| DLOZDRFLLIHYBD-UHFFFAOYSA-N | $8.9$ | | Wang et al. (2017) | Q | 80, 240 |
| MCM:H2C3OCNO3 | $9.6\times10^{2}$ | | Wang et al. (2017) | Q | 80, 238 |
| $C_4H_9NO_5$ | $9.1\times10^{2}$ | | Wang et al. (2017) | Q | 80, 239 |
| VJBSYBJCULHXOZ-UHFFFAOYSA-N | $3.0$ | | Wang et al. (2017) | Q | 80, 240 |
| MCM:PR2OHMONO3 | $1.1\times10^{3}$ | | Wang et al. (2017) | Q | 80, 238 |
| $C_4H_9NO_5$ | $3.8\times10^{2}$ | | Wang et al. (2017) | Q | 80, 239 |
| YOBYTHKQRGEURN-UHFFFAOYSA-N | $8.5\times10^{-1}$ | | Wang et al. (2017) | Q | 80, 240 |
| MCM:IEAPAN | $3.5\times10^{4}$ | | Wang et al. (2017) | Q | 80, 238 |
| $C_5H_7NO_7$ | $2.3\times10^{5}$ | | Wang et al. (2017) | Q | 80, 239 |
| NFJKKZSQGXUZSI-UHFFFAOYSA-N | $9.3\times10^{-1}$ | | Wang et al. (2017) | Q | 80, 240 |
| MCM:IECPAN | $3.5\times10^{4}$ | | Wang et al. (2017) | Q | 80, 238 |
| $C_5H_7NO_7$ | $9.1\times10^{4}$ | | Wang et al. (2017) | Q | 80, 239 |
| UMKIAKGFUATPHT-UHFFFAOYSA-N | $8.9\times10^{-1}$ | | Wang et al. (2017) | Q | 80, 240 |
| MCM:BOXEOHANO3 | $6.2\times10^{2}$ | | Wang et al. (2017) | Q | 80, 238 |
| $C_6H_{13}NO_5$ | $1.4\times10^{2}$ | | Wang et al. (2017) | Q | 80, 239 |
| NBMGDPHUZJHVTA-UHFFFAOYSA-N | $3.6$ | | Wang et al. (2017) | Q | 80, 240 |



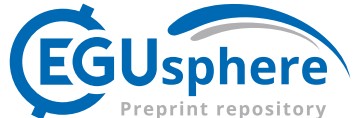

Table A4.6: Nitrates ($RONO_2$) (...continued)

| Substance Formula (Trivial Name) [CAS Registry Number] InChIKey | $H_s^{cp}$ (at $T^\ominus$) $\left[\dfrac{\text{mol}}{\text{m}^3\,\text{Pa}}\right]$ | $\dfrac{\text{d}\ln H_s^{cp}}{\text{d}(1/T)}$ [K] | Reference | Type | Note |
|---|---|---|---|---|---|
| MCM:BOXEOHBNO3 | $6.9\times10^2$ | | Wang et al. (2017) | Q | 80, 238 |
| $C_6H_{13}NO_5$ | $1.4\times10^2$ | | Wang et al. (2017) | Q | 80, 239 |
| YIIBDLSBOAJITL-UHFFFAOYSA-N | $3.8\times10^{-1}$ | | Wang et al. (2017) | Q | 80, 240 |
| MCM:BOXPOLANO3 | $5.8\times10^2$ | | Wang et al. (2017) | Q | 80, 238 |
| $C_7H_{15}NO_5$ | $8.3\times10^1$ | | Wang et al. (2017) | Q | 80, 239 |
| QDPFCROPCZKQGS-UHFFFAOYSA-N | 1.6 | | Wang et al. (2017) | Q | 80, 240 |
| MCM:BOXPOLBNO3 | $6.5\times10^2$ | | Wang et al. (2017) | Q | 80, 238 |
| $C_7H_{15}NO_5$ | $9.1\times10^1$ | | Wang et al. (2017) | Q | 80, 239 |
| UNWFJVCQCSAFHA-UHFFFAOYSA-N | $2.0\times10^{-1}$ | | Wang et al. (2017) | Q | 80, 240 |
| MCM:BCSOZNO3 | $1.3\times10^6$ | | Wang et al. (2017) | Q | 80, 238 |
| $C_{15}H_{25}NO_7$ | $5.3\times10^3$ | | Wang et al. (2017) | Q | 80, 239 |
| HGUJSQZJFHDFKY-UHFFFAOYSA-N | $1.1\times10^4$ | | Wang et al. (2017) | Q | 80, 240 |
| MCM:EPXDLPAN | $3.1\times10^5$ | | Wang et al. (2017) | Q | 80, 238 |
| $C_4H_3NO_7$ | $1.2\times10^5$ | | Wang et al. (2017) | Q | 80, 239 |
| VASNFLPKLUXZKF-UHFFFAOYSA-N | $8.3\times10^{-2}$ | | Wang et al. (2017) | Q | 80, 240 |
| MCM:EPXMDLPAN | $1.8\times10^5$ | | Wang et al. (2017) | Q | 80, 238 |
| $C_5H_5NO_7$ | $2.3\times10^4$ | | Wang et al. (2017) | Q | 80, 239 |
| KZASOTCYMWAOFE-UHFFFAOYSA-N | $3.5\times10^{-2}$ | | Wang et al. (2017) | Q | 80, 240 |
| MCM:BZEMUCPAN | $9.1\times10^5$ | | Wang et al. (2017) | Q | 80, 238 |
| $C_6H_5NO_7$ | $1.0\times10^5$ | | Wang et al. (2017) | Q | 80, 239 |
| RFFCGVCNFANFMQ-UHFFFAOYSA-N | 7.1 | | Wang et al. (2017) | Q | 80, 240 |
| MCM:EPXM2DLPAN | $9.6\times10^4$ | | Wang et al. (2017) | Q | 80, 238 |
| $C_6H_7NO_7$ | $6.3\times10^3$ | | Wang et al. (2017) | Q | 80, 239 |
| KYBHJMDXHJGYTK-UHFFFAOYSA-N | $4.2\times10^{-2}$ | | Wang et al. (2017) | Q | 80, 240 |
| MCM:EPXMEDLPAN | $8.0\times10^4$ | | Wang et al. (2017) | Q | 80, 238 |
| $C_7H_9NO_7$ | $3.5\times10^3$ | | Wang et al. (2017) | Q | 80, 239 |
| XUUSESMFYTYXHU-UHFFFAOYSA-N | $4.4\times10^{-2}$ | | Wang et al. (2017) | Q | 80, 240 |
| MCM:OXYMUCPAN | $2.5\times10^5$ | | Wang et al. (2017) | Q | 80, 238 |
| $C_8H_9NO_7$ | $7.1\times10^3$ | | Wang et al. (2017) | Q | 80, 239 |
| VIRSPYOLFRMIKR-UHFFFAOYSA-N | $1.1\times10^1$ | | Wang et al. (2017) | Q | 80, 240 |
| MCM:OETLMUCPAN | $2.2\times10^5$ | | Wang et al. (2017) | Q | 80, 238 |
| $C_9H_{11}NO_7$ | $4.3\times10^3$ | | Wang et al. (2017) | Q | 80, 239 |
| PYAKBDFKVOPHPS-UHFFFAOYSA-N | 2.8 | | Wang et al. (2017) | Q | 80, 240 |
| MCM:BZEMUCNO3 | $1.1\times10^8$ | | Wang et al. (2017) | Q | 80, 238 |
| $C_6H_7NO_7$ | $1.7\times10^8$ | | Wang et al. (2017) | Q | 80, 239 |
| UGJSRYKPNYMSMJ-UHFFFAOYSA-N | $1.7\times10^3$ | | Wang et al. (2017) | Q | 80, 240 |
| MCM:OXYMUCNO3 | $7.8\times10^8$ | | Wang et al. (2017) | Q | 80, 238 |
| $C_8H_{11}NO_7$ | $3.2\times10^7$ | | Wang et al. (2017) | Q | 80, 239 |
| PMXJWIAVDRSVOJ-UHFFFAOYSA-N | $2.6\times10^3$ | | Wang et al. (2017) | Q | 80, 240 |





Table A4.6: Nitrates ($RONO_2$) (...continued)

| Substance<br>Formula<br>(Trivial Name)<br>[CAS Registry Number]<br>InChIKey | $H_s^{cp}$ (at $T^\ominus$) $\left[\dfrac{\text{mol}}{\text{m}^3\,\text{Pa}}\right]$ | $\dfrac{\text{d}\ln H_s^{cp}}{\text{d}(1/T)}$ [K] | Reference | Type | Note |
|---|---|---|---|---|---|
| MCM:OETLMUCNO3<br>$C_9H_{13}NO_7$<br>KUQUSIKCNRTUGT-UHFFFAOYSA-N | $6.3\times10^8$<br>$1.9\times10^7$<br>$7.6\times10^2$ | | Wang et al. (2017)<br>Wang et al. (2017)<br>Wang et al. (2017) | Q<br>Q<br>Q | 80, 238<br>80, 239<br>80, 240 |
| MCM:EPXKTMPAN<br>$C_6H_7NO_7$<br>DBXZZDVADSLSOS-UHFFFAOYSA-N | $1.1\times10^5$<br>$2.6\times10^4$<br>$1.0\times10^{-1}$ | | Wang et al. (2017)<br>Wang et al. (2017)<br>Wang et al. (2017) | Q<br>Q<br>Q | 80, 238<br>80, 239<br>80, 240 |
| MCM:EPXMKTPAN<br>$C_6H_7NO_7$<br>UTKXQAWEMGMHOV-UHFFFAOYSA-N | $1.1\times10^5$<br>$2.6\times10^4$<br>$4.9\times10^{-1}$ | | Wang et al. (2017)<br>Wang et al. (2017)<br>Wang et al. (2017) | Q<br>Q<br>Q | 80, 238<br>80, 239<br>80, 240 |
| MCM:BOXPROANO3<br>$C_7H_{13}NO_5$<br>FWQXAWHLSYKKGA-UHFFFAOYSA-N | $1.0\times10^2$<br>$3.3\times10^1$<br>$1.1$ | | Wang et al. (2017)<br>Wang et al. (2017)<br>Wang et al. (2017) | Q<br>Q<br>Q | 80, 238<br>80, 239<br>80, 240 |
| MCM:EPXEKTPAN<br>$C_7H_9NO_7$<br>ZMAMXAXJQIDKIS-UHFFFAOYSA-N | $9.3\times10^4$<br>$1.6\times10^4$<br>$3.1\times10^{-1}$ | | Wang et al. (2017)<br>Wang et al. (2017)<br>Wang et al. (2017) | Q<br>Q<br>Q | 80, 238<br>80, 239<br>80, 240 |
| MCM:TLEMUCPAN<br>$C_7H_7NO_7$<br>AUCFXVOKAUAJMN-UHFFFAOYSA-N | $5.5\times10^5$<br>$2.0\times10^5$<br>$2.9\times10^1$ | | Wang et al. (2017)<br>Wang et al. (2017)<br>Wang et al. (2017) | Q<br>Q<br>Q | 80, 238<br>80, 239<br>80, 240 |
| MCM:EBZMUCPAN<br>$C_8H_9NO_7$<br>IJCUBYYAOSTAFY-UHFFFAOYSA-N | $4.9\times10^5$<br>$1.0\times10^5$<br>$9.8$ | | Wang et al. (2017)<br>Wang et al. (2017)<br>Wang et al. (2017) | Q<br>Q<br>Q | 80, 238<br>80, 239<br>80, 240 |
| MCM:MXYMUCPAN<br>$C_8H_9NO_7$<br>FZEVZMXGRRYMCK-UHFFFAOYSA-N | $3.0\times10^5$<br>$4.0\times10^4$<br>$1.2\times10^1$ | | Wang et al. (2017)<br>Wang et al. (2017)<br>Wang et al. (2017) | Q<br>Q<br>Q | 80, 238<br>80, 239<br>80, 240 |
| MCM:PXYMUCPAN<br>$C_8H_9NO_7$<br>JEHOQLADMHJYMP-UHFFFAOYSA-N | $3.0\times10^5$<br>$5.9\times10^4$<br>$8.1$ | | Wang et al. (2017)<br>Wang et al. (2017)<br>Wang et al. (2017) | Q<br>Q<br>Q | 80, 238<br>80, 239<br>80, 240 |
| MCM:IPBZMUCPAN<br>$C_9H_{11}NO_7$<br>JAYKUYZEOSXCLM-UHFFFAOYSA-N | $4.6\times10^5$<br>$6.2\times10^4$<br>$3.4$ | | Wang et al. (2017)<br>Wang et al. (2017)<br>Wang et al. (2017) | Q<br>Q<br>Q | 80, 238<br>80, 239<br>80, 240 |
| MCM:METLMUCPAN<br>$C_9H_{11}NO_7$<br>ZBESLNAUTTYMKY-UHFFFAOYSA-N | $2.8\times10^5$<br>$1.9\times10^4$<br>$5.1$ | | Wang et al. (2017)<br>Wang et al. (2017)<br>Wang et al. (2017) | Q<br>Q<br>Q | 80, 238<br>80, 239<br>80, 240 |
| MCM:PBZMUCPAN<br>$C_9H_{11}NO_7$<br>BNKFVONIJIEPHK-UHFFFAOYSA-N | $4.0\times10^5$<br>$6.0\times10^4$<br>$6.3$ | | Wang et al. (2017)<br>Wang et al. (2017)<br>Wang et al. (2017) | Q<br>Q<br>Q | 80, 238<br>80, 239<br>80, 240 |
| MCM:PETLMUCPAN<br>$C_9H_{11}NO_7$<br>ZDDBRQJYUBFFIJ-UHFFFAOYSA-N | $2.8\times10^5$<br>$2.8\times10^4$<br>$8.1$ | | Wang et al. (2017)<br>Wang et al. (2017)<br>Wang et al. (2017) | Q<br>Q<br>Q | 80, 238<br>80, 239<br>80, 240 |





Table A4.6: Nitrates ($RONO_2$) (...continued)

| Substance Formula (Trivial Name) [CAS Registry Number] InChIKey | $H_s^{cp}$ (at $T^{\ominus}$) $\left[\dfrac{\text{mol}}{\text{m}^3\,\text{Pa}}\right]$ | $\dfrac{\text{d}\ln H_s^{cp}}{\text{d}(1/T)}$ [K] | Reference | Type | Note |
|---|---|---|---|---|---|
| MCM:TM124MUPAN | $1.7\times10^5$ | | Wang et al. (2017) | Q | 80, 238 |
| $C_9H_{11}NO_7$ | $1.3\times10^4$ | | Wang et al. (2017) | Q | 80, 239 |
| TXJRVENADFPPLH-UHFFFAOYSA-N | $1.4\times10^1$ | | Wang et al. (2017) | Q | 80, 240 |
| MCM:TM135MUPAN | $2.0\times10^5$ | | Wang et al. (2017) | Q | 80, 238 |
| $C_9H_{11}NO_7$ | $3.7\times10^4$ | | Wang et al. (2017) | Q | 80, 239 |
| APWJKVMWYCHHEO-UHFFFAOYSA-N | 2.9 | | Wang et al. (2017) | Q | 80, 240 |
| MCM:DMEBMUPAN | $1.9\times10^5$ | | Wang et al. (2017) | Q | 80, 238 |
| $C_{10}H_{13}NO_7$ | $1.9\times10^4$ | | Wang et al. (2017) | Q | 80, 239 |
| OJIPANRKTQADST-UHFFFAOYSA-N | 2.1 | | Wang et al. (2017) | Q | 80, 240 |
| MCM:DETLMUPAN | $1.5\times10^5$ | | Wang et al. (2017) | Q | 80, 238 |
| $C_{11}H_{15}NO_7$ | $1.3\times10^4$ | | Wang et al. (2017) | Q | 80, 239 |
| WGGPAZKPQBFIGY-UHFFFAOYSA-N | 1.7 | | Wang et al. (2017) | Q | 80, 240 |
| MCM:TM123MUNO3 | $2.8\times10^7$ | | Wang et al. (2017) | Q | 80, 238 |
| $C_9H_{13}NO_7$ | $3.0\times10^7$ | | Wang et al. (2017) | Q | 80, 239 |
| CNKHLWZBBXJTJR-UHFFFAOYSA-N | $5.0\times10^2$ | | Wang et al. (2017) | Q | 80, 240 |
| MCM:TLEMUCNO3 | $1.5\times10^9$ | | Wang et al. (2017) | Q | 80, 238 |
| $C_7H_9NO_7$ | $3.5\times10^8$ | | Wang et al. (2017) | Q | 80, 239 |
| KLAQKLFQZHHGRD-UHFFFAOYSA-N | $8.3\times10^3$ | | Wang et al. (2017) | Q | 80, 240 |
| MCM:EBZMUCNO3 | $1.4\times10^9$ | | Wang et al. (2017) | Q | 80, 238 |
| $C_8H_{11}NO_7$ | $2.0\times10^8$ | | Wang et al. (2017) | Q | 80, 239 |
| ZZJNOSBMORNYPY-UHFFFAOYSA-N | $1.7\times10^4$ | | Wang et al. (2017) | Q | 80, 240 |
| MCM:MXYMUCNO3 | $8.5\times10^8$ | | Wang et al. (2017) | Q | 80, 238 |
| $C_8H_{11}NO_7$ | $7.8\times10^7$ | | Wang et al. (2017) | Q | 80, 239 |
| YZJRXCREGHPQLR-UHFFFAOYSA-N | $6.5\times10^3$ | | Wang et al. (2017) | Q | 80, 240 |
| MCM:PXYMUCNO3 | $8.5\times10^8$ | | Wang et al. (2017) | Q | 80, 238 |
| $C_8H_{11}NO_7$ | $9.3\times10^7$ | | Wang et al. (2017) | Q | 80, 239 |
| GICKPQRZPMBHKH-UHFFFAOYSA-N | $1.6\times10^3$ | | Wang et al. (2017) | Q | 80, 240 |
| MCM:IPBZMUCNO3 | $1.3\times10^9$ | | Wang et al. (2017) | Q | 80, 238 |
| $C_9H_{13}NO_7$ | $1.5\times10^8$ | | Wang et al. (2017) | Q | 80, 239 |
| UHFJHZLCDDVXBJ-UHFFFAOYSA-N | $1.1\times10^4$ | | Wang et al. (2017) | Q | 80, 240 |
| MCM:METLMUCNO3 | $7.6\times10^8$ | | Wang et al. (2017) | Q | 80, 238 |
| $C_9H_{13}NO_7$ | $4.6\times10^7$ | | Wang et al. (2017) | Q | 80, 239 |
| WWNOXWKNSLKIMY-UHFFFAOYSA-N | $9.8\times10^2$ | | Wang et al. (2017) | Q | 80, 240 |
| MCM:PBZMUCNO3 | $1.1\times10^9$ | | Wang et al. (2017) | Q | 80, 238 |
| $C_9H_{13}NO_7$ | $1.4\times10^8$ | | Wang et al. (2017) | Q | 80, 239 |
| QLDJRFHCOXXYTH-UHFFFAOYSA-N | $9.3\times10^3$ | | Wang et al. (2017) | Q | 80, 240 |
| MCM:PETLMUCNO3 | $7.6\times10^8$ | | Wang et al. (2017) | Q | 80, 238 |
| $C_9H_{13}NO_7$ | $5.8\times10^7$ | | Wang et al. (2017) | Q | 80, 239 |
| PIVPLJYVWKBKLM-UHFFFAOYSA-N | $1.0\times10^3$ | | Wang et al. (2017) | Q | 80, 240 |





Table A4.6: Nitrates ($RONO_2$) (...continued)

| Substance Formula (Trivial Name) [CAS Registry Number] InChIKey | $H_s^{cp}$ (at $T^\ominus$) $\left[\dfrac{\text{mol}}{\text{m}^3\,\text{Pa}}\right]$ | $\dfrac{\text{d}\ln H_s^{cp}}{\text{d}(1/T)}$ [K] | Reference | Type | Note |
|---|---|---|---|---|---|
| MCM:TM124MUNO3 $C_9H_{13}NO_7$ NIRISDMZVMYRLC-UHFFFAOYSA-N | $2.2\times10^7$ $7.8\times10^6$ $5.0\times10^1$ | | Wang et al. (2017) Wang et al. (2017) Wang et al. (2017) | Q Q Q | 80, 238 80, 239 80, 240 |
| MCM:TM135MUNO3 $C_9H_{13}NO_7$ WYJKGTDXAIWZNI-UHFFFAOYSA-N | $2.2\times10^7$ $1.4\times10^7$ $7.3\times10^1$ | | Wang et al. (2017) Wang et al. (2017) Wang et al. (2017) | Q Q Q | 80, 238 80, 239 80, 240 |
| MCM:DMEBMUNO3 $C_{10}H_{15}NO_7$ KKQBKVQGVIRRAG-UHFFFAOYSA-N | $1.9\times10^7$ $7.1\times10^6$ $2.9\times10^1$ | | Wang et al. (2017) Wang et al. (2017) Wang et al. (2017) | Q Q Q | 80, 238 80, 239 80, 240 |
| MCM:DETLMUNO3 $C_{11}H_{17}NO_7$ YPPYKYPDSBIFHG-UHFFFAOYSA-N | $1.4\times10^7$ $4.6\times10^6$ $2.8\times10^1$ | | Wang et al. (2017) Wang et al. (2017) Wang et al. (2017) | Q Q Q | 80, 238 80, 239 80, 240 |
| MCM:DMCNO3 $C_3H_5NO_6$ XNSIPMZBSRCICM-UHFFFAOYSA-N | $2.9\times10^1$ $3.6\times10^1$ $5.4\times10^{-2}$ | | Wang et al. (2017) Wang et al. (2017) Wang et al. (2017) | Q Q Q | 80, 238 80, 239 80, 240 |
| MCM:MMFNO3 $C_3H_5NO_6$ KROMLYNKDFILIM-UHFFFAOYSA-N | $1.5\times10^2$ $4.5$ $4.5\times10^{-2}$ | | Wang et al. (2017) Wang et al. (2017) Wang et al. (2017) | Q Q Q | 80, 238 80, 239 80, 240 |
| ISOP1N5OOH $C_5H_9NO_6$ LLNBZHMZWYQLAS-UHFFFAOYSA-N | $2.3\times10^7$ | 14000 | Wieser et al. (2023) | Q | 437 |
| C520ONO2 $C_5H_9NO_8$ ZSADVAXZKGJKSG-UHFFFAOYSA-N | $2.9\times10^9$ | 21000 | Wieser et al. (2023) | Q | 437 |
| ROO6R6ONO2 $C_6H_{11}NO_6$ CRECMHNOVVIMCR-UHFFFAOYSA-N | $6.1\times10^5$ | 12000 | Wieser et al. (2023) | Q | 437 |
| C624ONO2 $C_6H_{11}NO_7$ XYBUKFJLXVIRLY-UHFFFAOYSA-N | $2.0\times10^{10}$ | 19000 | Wieser et al. (2023) | Q | 437 |
| ROO6R5ONO2 $C_7H_{11}NO_7$ LNWOWXJMUJHMJJ-UHFFFAOYSA-N | $1.7\times10^5$ | 15000 | Wieser et al. (2023) | Q | 437 |
| ROO6R1ONO2 $C_{10}H_{17}NO_6$ JOFCABYMPNXMIS-UHFFFAOYSA-N | $1.4\times10^2$ | 15000 | Wieser et al. (2023) | Q | 437 |
| LIMAB15ONO2OOH $C_{10}H_{19}NO_7$ JBKULLYZCMWERK-UHFFFAOYSA-N | $5.0\times10^9$ | 19000 | Wieser et al. (2023) | Q | 437 |



### A4.7 Nitriles with oxygen (C, H, O, N)

Table A4.7: Nitriles with oxygen (C, H, O, N)

| Substance<br>Formula<br>(Trivial Name)<br>[CAS Registry Number]<br>InChIKey | $H_s^{cp}$<br>(at $T^{\ominus}$)<br>$\left[\dfrac{\text{mol}}{\text{m}^3\,\text{Pa}}\right]$ | $\dfrac{\mathrm{d}\ln H_s^{cp}}{\mathrm{d}(1/T)}$<br><br>[K] | Reference | Type | Note |
|---|---|---|---|---|---|
| isocyanic acid<br>HNCO<br>[75-13-8]<br>OWIKHYCFFJSOEH-UHFFFAOYSA-N | $2.3\times10^{-1}$<br>$2.6\times10^{-1}$<br>$2.1\times10^{-1}$ | 4700<br>4100 | Roberts and Liu (2019)<br>Borduas et al. (2016)<br>Roberts et al. (2011)<br>Burkholder et al. (2019) | M<br>M<br>M<br>W | 590<br>591<br><br>592 |
| methyl isocyanate<br>$CH_3NCO$<br>[624-83-9]<br>HAMGRBXTJNITHG-UHFFFAOYSA-N | $1.3\times10^{-2}$ | | Roberts and Liu (2019) | M | |
| hydroxyacetonitrile<br>$C_2H_3NO$<br>(glycolonitrile)<br>[107-16-4]<br>LTYRAPJYLUPLCI-UHFFFAOYSA-N | 1.3 | | HSDB (2015) | Q | 99 |
| 2-hydroxypropanenitrile<br>$C_3H_5NO$<br>[78-97-7]<br>WOFDVDFSGLBFAC-UHFFFAOYSA-N | 1.0 | | HSDB (2015) | Q | 99 |
| 3-hydroxypropanenitrile<br>$C_3H_5NO$<br>(ethylene cyanohydrin)<br>[109-78-4]<br>WSGYTJNNHPZFKR-UHFFFAOYSA-N | $1.3\times10^{3}$<br>$2.3\times10^{4}$<br>$3.2\times10^{2}$ | | Duchowicz et al. (2020)<br>HSDB (2015)<br>Duchowicz et al. (2020) | V<br>V<br>Q | 186 |
| methyl cyanoacetate<br>$C_4H_5NO_2$<br>[105-34-0]<br>ANGDWNBGPBMQHW-UHFFFAOYSA-N | $3.5\times10^{1}$ | | Ebert et al. (2023) | ? | 316 |
| cyanoethanoic acid, ethyl ester<br>$C_5H_7NO_2$<br>(ethyl cyanoacetate)<br>[105-56-6]<br>ZIUSEGSNTOUIPT-UHFFFAOYSA-N | $3.4\times10^{1}$<br>$3.4\times10^{1}$<br>$1.3\times10^{1}$<br>$7.7\times10^{1}$<br>4.9<br>7.7<br>$3.5\times10^{1}$ | | Duchowicz et al. (2020)<br>HSDB (2015)<br>Duchowicz et al. (2020)<br>Hilal et al. (2008)<br>Modarresi et al. (2007)<br>Yao et al. (2002)<br>Yaws (1999) | V<br>V<br>Q<br>Q<br>Q<br>Q<br>? | 186<br><br><br><br>67<br>229<br>21 |
| 2-hydroxybenzoic acid nitrile<br>$C_7H_5NO$<br>(2-cyanophenol)<br>[611-20-1]<br>CHZCERSEMVWNHL-UHFFFAOYSA-N | $2.8\times10^{1}$ | | Hilal et al. (2008) | Q | |





Table A4.7: Nitriles with oxygen (C, H, O, N) (...continued)

| Substance Formula (Trivial Name) [CAS Registry Number] InChIKey | $H_s^{cp}$ (at $T^{\ominus}$) $\left[\dfrac{\text{mol}}{\text{m}^3\,\text{Pa}}\right]$ | $\dfrac{\text{d}\ln H_s^{cp}}{\text{d}(1/T)}$ [K] | Reference | Type | Note |
|---|---|---|---|---|---|
| 3-hydroxybenzoic acid nitrile $C_7H_5NO$ (3-cyanophenol) [873-62-1] SGHBRHKBCLLVCI-UHFFFAOYSA-N | $4.0\times10^4$ $1.6\times10^3$ $3.6\times10^3$ $3.3\times10^5$ $3.8\times10^3$ | | Hilal et al. (2008) Modarresi et al. (2007) English and Carroll (2001) Nirmalakhandan et al. (1997) Abraham et al. (1990) | Q Q Q Q ? | 67 230, 231 |
| 4-hydroxybenzoic acid nitrile $C_7H_5NO$ (4-cyanophenol) [767-00-0] CVNOWLNNPYYEOH-UHFFFAOYSA-N | $1.4\times10^4$ $2.0\times10^3$ $3.3\times10^5$ $1.2\times10^4$ | | Hilal et al. (2008) Modarresi et al. (2007) Nirmalakhandan et al. (1997) Abraham et al. (1990) | Q Q Q ? | 67 |
| phenyl isocyanate $C_7H_5NO$ [103-71-9] DGTNSSLYPYDJGL-UHFFFAOYSA-N | $2.5\times10^{-5}$ $2.5\times10^{-5}$ $1.8\times10^{-5}$ | | Yaws (2003) Gharagheizi et al. (2010) Yaws (1999) | X Q ? | 237, 12 246 21, 12 |
| 1,1',1''-nitrilotris-2-propanol $C_9H_{21}NO_3$ (triisopropanolamine) [122-20-3] SLINHMUFWFWBMU-UHFFFAOYSA-N | $1.0\times10^6$ | | HSDB (2015) | Q | 447 |
| cyometrinil $C_{10}H_7N_3O$ [78370-21-5] PYKLUAIDKVVEOS-JLHYYAGUSA-N | $1.1\times10^4$ | | MacBean (2012a) | ? | |
| fenpropathrin $C_{22}H_{23}NO_3$ [39515-41-8] XQUXKZZNEFRCAW-UHFFFAOYSA-N | $5.5\times10^{-2}$ $1.7\times10^1$ | | HSDB (2015) Siebers and Mattusch (1996) | V V | 12 |





### A4.8 Nitro compounds (RNO$_2$)

Table A4.8: Nitro compounds (RNO$_2$)

| Substance Formula (Trivial Name) [CAS Registry Number] InChIKey | $H_s^{cp}$ (at $T^\ominus$) $\left[\dfrac{\text{mol}}{\text{m}^3\,\text{Pa}}\right]$ | $\dfrac{\text{d}\ln H_s^{cp}}{\text{d}(1/T)}$ [K] | Reference | Type | Note |
|---|---|---|---|---|---|
| nitromethane | $3.4\times10^{-1}$ | 4000 | Burkholder et al. (2019) | L | |
| CH$_3$NO$_2$ | $3.4\times10^{-1}$ | 4000 | Burkholder et al. (2015) | L | |
| [75-52-5] | $3.6\times10^{-1}$ | 3900 | Brockbank (2013) | L | 1 |
| LYGJENNIWJXYER-UHFFFAOYSA-N | $3.4\times10^{-1}$ | 4000 | Sander et al. (2011) | L | |
| | $3.4\times10^{-1}$ | 4000 | Sander et al. (2006) | L | |
| | $3.5\times10^{-1}$ | 4000 | Beneš and Dohnal (1999) | M | |
| | $3.6\times10^{-1}$ | | Park et al. (1987) | M | |
| | $4.5\times10^{-1}$ | | Rohrschneider (1973) | M | |
| | $3.4\times10^{-2}$ | | Yaws (2003) | X | 237 |
| | $3.5\times10^{-1}$ | | Gaffney and Senum (1984) | X | 389 |
| | $4.4\times10^{-1}$ | | Hayer et al. (2022) | Q | 20 |
| | $2.5\times10^{-1}$ | | Keshavarz et al. (2022) | Q | |
| | $6.2\times10^{-1}$ | | Duchowicz et al. (2020) | Q | 184 |
| | $1.2\times10^{-1}$ | | Raventos-Duran et al. (2010) | Q | 242, 243 |
| | $2.0\times10^{-1}$ | | Raventos-Duran et al. (2010) | Q | 244 |
| | $2.5\times10^{-1}$ | | Raventos-Duran et al. (2010) | Q | 245 |
| | $3.3\times10^{-2}$ | | Gharagheizi et al. (2010) | Q | 246 |
| | $3.4\times10^{-1}$ | | Hilal et al. (2008) | Q | |
| | $3.3\times10^{-2}$ | | Modarresi et al. (2007) | Q | 67 |
| | | 3700 | Kühne et al. (2005) | Q | |
| | $2.3\times10^{-1}$ | | Yaffe et al. (2003) | Q | 248, 272 |
| | $6.2\times10^{-1}$ | | English and Carroll (2001) | Q | 230, 231 |
| | $2.8\times10^{-2}$ | | Katritzky et al. (1998) | Q | |
| | $7.3\times10^{-2}$ | | Nirmalakhandan et al. (1997) | Q | |
| | $3.5\times10^{-1}$ | | Duchowicz et al. (2020) | ? | 185, 21 |
| | | 3500 | Kühne et al. (2005) | ? | |
| | $3.4\times10^{-2}$ | | Yaws (1999) | ? | 21 |
| | $1.8\times10^{-1}$ | | Abraham and Weathersby (1994) | ? | 21 |
| | $3.6\times10^{-2}$ | | Yaws and Yang (1992) | ? | 21 |
| | $3.6\times10^{-1}$ | | Abraham et al. (1990) | ? | |
| nitromethane-13C CH$_3$NO$_2$ [32480-00-5] LYGJENNIWJXYER-OUBTZVSYSA-N | $4.8\times10^{-1}$ | 5000 | Hiatt (2013) | M | |
| nitroethane | $2.1\times10^{-1}$ | 4400 | Burkholder et al. (2019) | L | |
| C$_2$H$_5$NO$_2$ | $2.1\times10^{-1}$ | 4400 | Burkholder et al. (2015) | L | |
| [79-24-3] | $2.1\times10^{-1}$ | 4700 | Brockbank (2013) | L | 1 |
| MCSAJNNLRCFZED-UHFFFAOYSA-N | $2.1\times10^{-1}$ | 4400 | Sander et al. (2011) | L | |
| | $2.1\times10^{-1}$ | 4400 | Sander et al. (2006) | L | |
| | $2.2\times10^{-1}$ | 4400 | Beneš and Dohnal (1999) | M | |
| | $1.4\times10^{-1}$ | | Friant and Suffet (1979) | M | 38, 593 |
| | $1.9\times10^{-1}$ | | Hwang et al. (1992) | V | |
| | $2.1\times10^{-1}$ | | Hine and Mookerjee (1975) | V | |
| | $2.1\times10^{-1}$ | | Gaffney and Senum (1984) | X | 389 |



Table A4.8: Nitro compounds ($RNO_2$) (... continued)

| Substance Formula (Trivial Name) [CAS Registry Number] InChIKey | $H_s^{cp}$ (at $T^\ominus$) $\left[\dfrac{\text{mol}}{\text{m}^3\,\text{Pa}}\right]$ | $\dfrac{\text{d}\ln H_s^{cp}}{\text{d}(1/T)}$ [K] | Reference | Type | Note |
|---|---|---|---|---|---|
| | $3.4\times10^{-1}$ | | Keshavarz et al. (2022) | Q | |
| | $3.7\times10^{-1}$ | | Duchowicz et al. (2020) | Q | 184 |
| | $2.1\times10^{-1}$ | | Li et al. (2014) | Q | 241 |
| | $9.9\times10^{-2}$ | | Raventos-Duran et al. (2010) | Q | 242, 243 |
| | $2.0\times10^{-1}$ | | Raventos-Duran et al. (2010) | Q | 244 |
| | $1.6\times10^{-1}$ | | Raventos-Duran et al. (2010) | Q | 245 |
| | $2.2\times10^{-1}$ | | Hilal et al. (2008) | Q | |
| | $3.0\times10^{-2}$ | | Modarresi et al. (2007) | Q | 67 |
| | | 4100 | Kühne et al. (2005) | Q | |
| | $2.3\times10^{-1}$ | | Yaffe et al. (2003) | Q | 248, 249 |
| | $3.1\times10^{-1}$ | | English and Carroll (2001) | Q | 230, 260 |
| | $2.6\times10^{-2}$ | | Katritzky et al. (1998) | Q | |
| | $6.1\times10^{-2}$ | | Nirmalakhandan et al. (1997) | Q | |
| | $1.5\times10^{-1}$ | | Suzuki et al. (1992) | Q | 232 |
| | $2.1\times10^{-1}$ | | Duchowicz et al. (2020) | ? | 185, 21 |
| | | 4200 | Kühne et al. (2005) | ? | |
| | $2.5\times10^{-1}$ | | Yaws (1999) | ? | 21 |
| | $2.1\times10^{-1}$ | | Abraham et al. (1990) | ? | |
| 1-nitropropane $C_3H_7NO_2$ [108-03-2] JSZOAYXJRCEYSX-UHFFFAOYSA-N | $1.3\times10^{-1}$ | 4700 | Burkholder et al. (2019) | L | |
| | $1.3\times10^{-1}$ | 4700 | Burkholder et al. (2015) | L | |
| | $1.4\times10^{-1}$ | 5100 | Brockbank (2013) | L | 1 |
| | $1.3\times10^{-1}$ | 4700 | Sander et al. (2011) | L | |
| | $1.3\times10^{-1}$ | 4700 | Sander et al. (2006) | L | |
| | $1.3\times10^{-1}$ | 4700 | Beneš and Dohnal (1999) | M | |
| | $1.6\times10^{-1}$ | | Welke et al. (1998) | M | |
| | $1.1\times10^{-1}$ | | Hine and Mookerjee (1975) | V | |
| | $9.2\times10^{-2}$ | | Keshavarz et al. (2022) | Q | |
| | $4.8\times10^{-1}$ | | Duchowicz et al. (2020) | Q | 299 |
| | $7.8\times10^{-2}$ | | Raventos-Duran et al. (2010) | Q | 242, 243 |
| | $9.9\times10^{-2}$ | | Raventos-Duran et al. (2010) | Q | 244 |
| | $1.2\times10^{-1}$ | | Raventos-Duran et al. (2010) | Q | 245 |
| | $1.5\times10^{-1}$ | | Hilal et al. (2008) | Q | |
| | $2.3\times10^{-2}$ | | Modarresi et al. (2007) | Q | 67 |
| | | 4400 | Kühne et al. (2005) | Q | |
| | $1.2\times10^{-1}$ | | Yaffe et al. (2003) | Q | 248, 249 |
| | $1.2\times10^{-1}$ | | English and Carroll (2001) | Q | 230, 231 |
| | $3.1\times10^{-2}$ | | Katritzky et al. (1998) | Q | |
| | $4.7\times10^{-2}$ | | Nirmalakhandan et al. (1997) | Q | |
| | $7.2\times10^{-3}$ | | Suzuki et al. (1992) | Q | 232 |
| | $1.1\times10^{-1}$ | | Duchowicz et al. (2020) | ? | 185, 21 |
| | | 4400 | Kühne et al. (2005) | ? | |
| | $1.2\times10^{-1}$ | | Yaws (1999) | ? | 21, 12 |
| | $1.6\times10^{-1}$ | | Yaws and Yang (1992) | ? | 21, 12 |
| | $1.1\times10^{-1}$ | | Abraham et al. (1990) | ? | |



Table A4.8: Nitro compounds ($RNO_2$) (...continued)

| Substance Formula (Trivial Name) [CAS Registry Number] InChIKey | $H_s^{cp}$ (at $T^\ominus$) $\left[\dfrac{\text{mol}}{\text{m}^3\,\text{Pa}}\right]$ | $\dfrac{\text{d}\ln H_s^{cp}}{\text{d}(1/T)}$ [K] | Reference | Type | Note |
|---|---|---|---|---|---|
| 2-nitropropane | $8.3\times10^{-2}$ | 4500 | Burkholder et al. (2019) | L | |
| $CH_3CH(NO_2)CH_3$ | $8.3\times10^{-2}$ | 4500 | Burkholder et al. (2015) | L | |
| [79-46-9] | $8.6\times10^{-2}$ | 4800 | Brockbank (2013) | L | 1 |
| FGLBSLMDCBOPQK-UHFFFAOYSA-N | $8.3\times10^{-2}$ | 4500 | Sander et al. (2011) | L | |
| | $8.3\times10^{-2}$ | 4500 | Sander et al. (2006) | L | |
| | $8.4\times10^{-2}$ | 4500 | Beneš and Dohnal (1999) | M | |
| | $8.3\times10^{-2}$ | | Duchowicz et al. (2020) | V | 186 |
| | $8.3\times10^{-2}$ | | HSDB (2015) | V | |
| | $8.0\times10^{-2}$ | | Hine and Mookerjee (1975) | V | |
| | $2.2\times10^{-1}$ | | Duchowicz et al. (2020) | Q | |
| | $7.8\times10^{-2}$ | | Raventos-Duran et al. (2010) | Q | 242, 243 |
| | $6.2\times10^{-2}$ | | Raventos-Duran et al. (2010) | Q | 244 |
| | $1.2\times10^{-1}$ | | Raventos-Duran et al. (2010) | Q | 245 |
| | $7.2\times10^{-2}$ | | Hilal et al. (2008) | Q | |
| | $2.7\times10^{-2}$ | | Modarresi et al. (2007) | Q | 67 |
| | | 4400 | Kühne et al. (2005) | Q | |
| | $8.6\times10^{-2}$ | | Yaffe et al. (2003) | Q | 248, 249 |
| | $7.2\times10^{-2}$ | | English and Carroll (2001) | Q | 230, 274 |
| | $2.1\times10^{-2}$ | | Katritzky et al. (1998) | Q | |
| | $4.1\times10^{-2}$ | | Nirmalakhandan et al. (1997) | Q | |
| | $8.4\times10^{-3}$ | | Suzuki et al. (1992) | Q | 232 |
| | | 4400 | Kühne et al. (2005) | ? | |
| | $8.2\times10^{-2}$ | | Yaws (1999) | ? | 21, 12 |
| | $1.1\times10^{-1}$ | | Yaws and Yang (1992) | ? | 21, 12 |
| | $8.0\times10^{-2}$ | | Abraham et al. (1990) | ? | |
| 1-nitrobutane | $7.6\times10^{-2}$ | | Brockbank (2013) | L | |
| $C_4H_9NO_2$ | $8.3\times10^{-2}$ | | Duchowicz et al. (2020) | V | 186 |
| [627-05-4] | $5.6\times10^{-1}$ | | Duchowicz et al. (2020) | Q | |
| NALZTFARIYUCBY-UHFFFAOYSA-N | $9.7\times10^{-2}$ | | Hilal et al. (2008) | Q | |
| | $1.8\times10^{-2}$ | | Modarresi et al. (2007) | Q | 67 |
| | $3.7\times10^{-2}$ | | Nirmalakhandan et al. (1997) | Q | |
| | $7.5\times10^{-2}$ | | Abraham et al. (1990) | ? | |
| *tert*-butylnitrite | $7.9\times10^{-3}$ | | Hilal et al. (2008) | Q | |
| $C_4H_9ONO$ | | | | | |
| [540-80-7] | | | | | |
| IOGXOCVLYRDXLW-UHFFFAOYSA-N | | | | | |
| 1-nitropentane | $4.7\times10^{-2}$ | | Amoore and Buttery (1978) | V | |
| $C_5H_{11}NO_2$ | $6.0\times10^{-2}$ | | Hilal et al. (2008) | Q | |
| [628-05-7] | $1.4\times10^{-2}$ | | Modarresi et al. (2007) | Q | 67 |
| BVALZCVRLDMXOQ-UHFFFAOYSA-N | $2.9\times10^{-2}$ | | Nirmalakhandan et al. (1997) | Q | |
| | $4.7\times10^{-2}$ | | Abraham et al. (1990) | ? | |



Table A4.8: Nitro compounds ($RNO_2$) (...continued)

| Substance<br>Formula<br>(Trivial Name)<br>[CAS Registry Number]<br>InChIKey | $H_s^{cp}$ (at $T^{\ominus}$) $\left[\dfrac{\text{mol}}{\text{m}^3\,\text{Pa}}\right]$ | $\dfrac{\mathrm{d}\ln H_s^{cp}}{\mathrm{d}(1/T)}$ [K] | Reference | Type | Note |
|---|---|---|---|---|---|
| tris(hydroxymethyl)ethane trinitrate | $2.2\times10^3$ | | Zhang et al. (2010) | Q | 287, 288 |
| $C_5H_9N_3O_9$ | $1.4\times10^2$ | | Zhang et al. (2010) | Q | 287, 289 |
| [3032-55-1] | $2.4\times10^3$ | | Zhang et al. (2010) | Q | 287, 290 |
| IPPYBNCEPZCLNI-UHFFFAOYSA-N | $3.4\times10^1$ | | Zhang et al. (2010) | Q | 287, 291 |
| 1-nitrohexane | $4.5\times10^{-2}$ | | Hilal et al. (2008) | Q | |
| $C_6H_{13}NO_2$ | | | | | |
| [646-14-0] | | | | | |
| FEYJIFXFOHFGCC-UHFFFAOYSA-N | | | | | |
| nitrocyclohexane | $2.4\times10^{-1}$ | | Hilal et al. (2008) | Q | |
| $C_6H_{11}NO_2$ | | | | | |
| [1122-60-7] | | | | | |
| NJNQUTDUIPVROZ-UHFFFAOYSA-N | | | | | |
| 2-nitroethanol | $2.0\times10^2$ | | Raventos-Duran et al. (2010) | Q | 242, 243 |
| $C_2H_5NO_3$ | $3.1\times10^2$ | | Raventos-Duran et al. (2010) | Q | 244 |
| [625-48-9] | $4.9\times10^3$ | | Raventos-Duran et al. (2010) | Q | 245 |
| KIPMDPDAFINLIV-UHFFFAOYSA-N | $1.6\times10^2$ | | Hilal et al. (2008) | Q | |
| 1-nitro-2-propanol | $1.2\times10^2$ | | Raventos-Duran et al. (2010) | Q | 242, 243 |
| $C_3H_7NO_3$ | $2.5\times10^2$ | | Raventos-Duran et al. (2010) | Q | 244 |
| [3156-73-8] | $3.9\times10^3$ | | Raventos-Duran et al. (2010) | Q | 245 |
| PFNCKQIYLAVYJF-UHFFFAOYSA-N | $7.9\times10^1$ | | Hilal et al. (2008) | Q | |
| 2-nitro-1-propanol | $1.2\times10^2$ | | Raventos-Duran et al. (2010) | Q | 242, 243 |
| $C_3H_7NO_3$ | $2.5\times10^2$ | | Raventos-Duran et al. (2010) | Q | 244 |
| [2902-96-7] | $3.9\times10^3$ | | Raventos-Duran et al. (2010) | Q | 245 |
| PCNWBUOSTLGPMI-UHFFFAOYSA-N | $9.9\times10^1$ | | Hilal et al. (2008) | Q | |
| 1-nitro-2-butanol | $7.3\times10^1$ | | Hilal et al. (2008) | Q | |
| $C_4H_9NO_3$ | | | | | |
| [3156-74-9] | | | | | |
| FMEFHKJRIGHSLB-UHFFFAOYSA-N | | | | | |
| 2-nitro-1-butanol | $9.9\times10^1$ | | Raventos-Duran et al. (2010) | Q | 271, 243 |
| $C_4H_9NO_3$ | $2.0\times10^2$ | | Raventos-Duran et al. (2010) | Q | 244 |
| [609-31-4] | $2.5\times10^3$ | | Raventos-Duran et al. (2010) | Q | 245 |
| MHIHRIPETCJEMQ-UHFFFAOYSA-N | $7.5\times10^1$ | | Hilal et al. (2008) | Q | |
| 3-nitro-2-butanol | $9.9\times10^1$ | | Raventos-Duran et al. (2010) | Q | 242, 243 |
| $C_4H_9NO_3$ | $2.0\times10^2$ | | Raventos-Duran et al. (2010) | Q | 244 |
| [6270-16-2] | $2.5\times10^3$ | | Raventos-Duran et al. (2010) | Q | 245 |
| OJVOGABFNZDOOZ-UHFFFAOYSA-N | $5.7\times10^1$ | | Hilal et al. (2008) | Q | |
| nitroguanidine | $2.2\times10^{10}$ | | HSDB (2015) | V | |
| $CH_4N_4O_2$ | | | | | |
| [556-88-7] | | | | | |
| IDCPFAYURAQKDZ-UHFFFAOYSA-N | | | | | |




Table A4.8: Nitro compounds ($RNO_2$) (...continued)

| Substance Formula (Trivial Name) [CAS Registry Number] InChIKey | $H_s^{cp}$ (at $T^{\ominus}$) $\left[\dfrac{\mathrm{mol}}{\mathrm{m^3\,Pa}}\right]$ | $\dfrac{\mathrm{d}\ln H_s^{cp}}{\mathrm{d}(1/T)}$ [K] | Reference | Type | Note |
|---|---|---|---|---|---|
| tetranitromethane $CN_4O_8$ [509-14-8] NYTOUQBROMCLBJ-UHFFFAOYSA-N | $4.1\times10^{-3}$ | | HSDB (2015) | V | |
| N-methyl-N'-nitro-N-nitrosoguanidine $C_2H_5N_5O_3$ [70-25-7] VZUNGTLZRAYYDE-UHFFFAOYSA-N | $8.2\times10^{6}$ | | HSDB (2015) | Q | 99 |
| 2-(hydroxymethyl)-2-nitro-1,3-propanediol $C_4H_9NO_5$ [126-11-4] OLQJQHSAWMFDJE-UHFFFAOYSA-N | $2.1\times10^{6}$ | | HSDB (2015) | Q | 99 |
| MCM:NC4DCO2H $C_4H_3NO_5$ ODFCIXRXNCCVGH-UHFFFAOYSA-N | $6.9\times10^{6}$ $3.5\times10^{6}$ $2.3\times10^{3}$ | | Wang et al. (2017) Wang et al. (2017) Wang et al. (2017) | Q Q Q | 80, 238 80, 239 80, 240 |
| MCM:NC4MDCO2H $C_5H_5NO_5$ LXEJKNKBIPISQO-UHFFFAOYSA-N | $4.2\times10^{6}$ $4.7\times10^{6}$ $6.5\times10^{3}$ | | Wang et al. (2017) Wang et al. (2017) Wang et al. (2017) | Q Q Q | 80, 238 80, 239 80, 240 |
| MCM:NC4EDCO2H $C_6H_7NO_5$ GVSLZEQKUBDLJE-UHFFFAOYSA-N | $3.6\times10^{6}$ $6.9\times10^{6}$ $3.9\times10^{3}$ | | Wang et al. (2017) Wang et al. (2017) Wang et al. (2017) | Q Q Q | 80, 238 80, 239 80, 240 |
| MCM:DNPHENOOH $C_6H_6N_2O_{10}$ AKJGOKAIBHUAAE-UHFFFAOYSA-N | $7.3\times10^{16}$ $8.0\times10^{10}$ $5.4\times10^{7}$ | | Wang et al. (2017) Wang et al. (2017) Wang et al. (2017) | Q Q Q | 80, 238 80, 239 80, 240 |
| MCM:NCATECOOH $C_6H_7NO_9$ UERHMILVGXASDJ-UHFFFAOYSA-N | $8.1\times10^{17}$ $1.6\times10^{10}$ $4.4\times10^{9}$ | | Wang et al. (2017) Wang et al. (2017) Wang et al. (2017) | Q Q Q | 80, 238 80, 239 80, 240 |
| MCM:NC5MDCO2H $C_6H_7NO_5$ VRGZGZRSZACGSZ-UHFFFAOYSA-N | $2.8\times10^{6}$ $1.1\times10^{7}$ $2.5\times10^{3}$ | | Wang et al. (2017) Wang et al. (2017) Wang et al. (2017) | Q Q Q | 80, 238 80, 239 80, 240 |
| MCM:NC4IPDCO2H $C_7H_9NO_5$ VOBLVGUNJWYRHD-UHFFFAOYSA-N | $3.5\times10^{6}$ $3.5\times10^{6}$ $1.8\times10^{3}$ | | Wang et al. (2017) Wang et al. (2017) Wang et al. (2017) | Q Q Q | 80, 238 80, 239 80, 240 |
| MCM:NC4PDCO2H $C_7H_9NO_5$ MSTRQTMGALLSFO-UHFFFAOYSA-N | $3.0\times10^{6}$ $4.2\times10^{6}$ $2.2\times10^{3}$ | | Wang et al. (2017) Wang et al. (2017) Wang et al. (2017) | Q Q Q | 80, 238 80, 239 80, 240 |
| MCM:DNCRESOOH $C_7H_8N_2O_{10}$ FHWDGQSKNURQKZ-UHFFFAOYSA-N | $4.0\times10^{16}$ $3.6\times10^{10}$ $1.4\times10^{6}$ | | Wang et al. (2017) Wang et al. (2017) Wang et al. (2017) | Q Q Q | 80, 238 80, 239 80, 240 |



Table A4.8: Nitro compounds ($RNO_2$) (. . . continued)

| Substance Formula (Trivial Name) [CAS Registry Number] InChIKey | $H_s^{cp}$ (at $T^{\ominus}$) $\left[\dfrac{\mathrm{mol}}{\mathrm{m}^3\,\mathrm{Pa}}\right]$ | $\dfrac{\mathrm{d}\ln H_s^{cp}}{\mathrm{d}(1/T)}$ [K] | Reference | Type | Note |
|---|---|---|---|---|---|
| MCM:MNCATECOOH | $3.0\times10^{17}$ | | Wang et al. (2017) | Q | 80, 238 |
| $C_7H_9NO_9$ | $1.7\times10^{11}$ | | Wang et al. (2017) | Q | 80, 239 |
| OCHAAZKMUDWIOE-UHFFFAOYSA-N | $4.4\times10^{7}$ | | Wang et al. (2017) | Q | 80, 240 |
| MCM:TL4ONO2OOH | $1.4\times10^{14}$ | | Wang et al. (2017) | Q | 80, 238 |
| $C_7H_9NO_8$ | $2.8\times10^{9}$ | | Wang et al. (2017) | Q | 80, 239 |
| MEXRHSIIQRTIBR-UHFFFAOYSA-N | $3.5\times10^{6}$ | | Wang et al. (2017) | Q | 80, 240 |
| MCM:DNEBNZLOOH | $3.6\times10^{16}$ | | Wang et al. (2017) | Q | 80, 238 |
| $C_8H_{10}N_2O_{10}$ | $2.3\times10^{10}$ | | Wang et al. (2017) | Q | 80, 239 |
| FNCRENCSRXRIPJ-UHFFFAOYSA-N | $3.0\times10^{5}$ | | Wang et al. (2017) | Q | 80, 240 |
| MCM:DNMXYOLOOH | $2.7\times10^{16}$ | | Wang et al. (2017) | Q | 80, 238 |
| $C_8H_{10}N_2O_{10}$ | $2.2\times10^{11}$ | | Wang et al. (2017) | Q | 80, 239 |
| KSAVSEUSLRIGAB-UHFFFAOYSA-N | $5.9\times10^{7}$ | | Wang et al. (2017) | Q | 80, 240 |
| MCM:DNOXYOLOOH | $3.2\times10^{16}$ | | Wang et al. (2017) | Q | 80, 238 |
| $C_8H_{10}N_2O_{10}$ | $5.9\times10^{10}$ | | Wang et al. (2017) | Q | 80, 239 |
| PBZRLTNSSYSOGL-UHFFFAOYSA-N | $7.8\times10^{4}$ | | Wang et al. (2017) | Q | 80, 240 |
| MCM:DNPXYOLOOH | $2.7\times10^{16}$ | | Wang et al. (2017) | Q | 80, 238 |
| $C_8H_{10}N_2O_{10}$ | $3.1\times10^{10}$ | | Wang et al. (2017) | Q | 80, 239 |
| AYHFAKIJCOHXBG-UHFFFAOYSA-N | $1.5\times10^{5}$ | | Wang et al. (2017) | Q | 80, 240 |
| MCM:ENCATECOOH | $2.6\times10^{17}$ | | Wang et al. (2017) | Q | 80, 238 |
| $C_8H_{11}NO_9$ | $8.9\times10^{10}$ | | Wang et al. (2017) | Q | 80, 239 |
| QJBJEZDKVLRDGQ-UHFFFAOYSA-N | $1.8\times10^{7}$ | | Wang et al. (2017) | Q | 80, 240 |
| MCM:MXNCATCOOH | $1.6\times10^{17}$ | | Wang et al. (2017) | Q | 80, 238 |
| $C_8H_{11}NO_9$ | $6.2\times10^{10}$ | | Wang et al. (2017) | Q | 80, 239 |
| QYJOURKZCRGVGB-UHFFFAOYSA-N | $3.4\times10^{7}$ | | Wang et al. (2017) | Q | 80, 240 |
| MCM:MXOHNO2OOH | $1.8\times10^{14}$ | | Wang et al. (2017) | Q | 80, 238 |
| $C_8H_{11}NO_8$ | $1.2\times10^{9}$ | | Wang et al. (2017) | Q | 80, 239 |
| RRWTXIWIABDEOZ-UHFFFAOYSA-N | $1.3\times10^{4}$ | | Wang et al. (2017) | Q | 80, 240 |
| MCM:OXNCATCOOH | $3.9\times10^{17}$ | | Wang et al. (2017) | Q | 80, 238 |
| $C_8H_{11}NO_9$ | $1.2\times10^{10}$ | | Wang et al. (2017) | Q | 80, 239 |
| GWPIPVPJLOZVKC-UHFFFAOYSA-N | $3.7\times10^{5}$ | | Wang et al. (2017) | Q | 80, 240 |
| MCM:PXNCATCOOH | $3.0\times10^{17}$ | | Wang et al. (2017) | Q | 80, 238 |
| $C_8H_{11}NO_9$ | $5.5\times10^{9}$ | | Wang et al. (2017) | Q | 80, 239 |
| RMUFWTRDQQMPCD-UHFFFAOYSA-N | $1.4\times10^{7}$ | | Wang et al. (2017) | Q | 80, 240 |
| MCM:TM124NOOH | $1.8\times10^{14}$ | | Wang et al. (2017) | Q | 80, 238 |
| $C_8H_{11}NO_8$ | $1.3\times10^{9}$ | | Wang et al. (2017) | Q | 80, 239 |
| LGMZUZOZZXWRQX-UHFFFAOYSA-N | $5.1\times10^{4}$ | | Wang et al. (2017) | Q | 80, 240 |
| MCM:DNIPBZLOOH | $3.2\times10^{16}$ | | Wang et al. (2017) | Q | 80, 238 |
| $C_9H_{12}N_2O_{10}$ | $2.3\times10^{10}$ | | Wang et al. (2017) | Q | 80, 239 |
| YKBFACRSJWRBHR-UHFFFAOYSA-N | $3.7\times10^{6}$ | | Wang et al. (2017) | Q | 80, 240 |



Table A4.8: Nitro compounds ($RNO_2$) (...continued)

| Substance Formula (Trivial Name) [CAS Registry Number] InChIKey | $H_s^{cp}$ (at $T^\ominus$) $\left[\dfrac{\mathrm{mol}}{\mathrm{m^3\,Pa}}\right]$ | $\dfrac{\mathrm{d}\ln H_s^{cp}}{\mathrm{d}(1/T)}$ [K] | Reference | Type | Note |
|---|---|---|---|---|---|
| MCM:DNMETOLOOH | $2.1\times10^{16}$ | | Wang et al. (2017) | Q | 80, 238 |
| $C_9H_{12}N_2O_{10}$ | $1.4\times10^{11}$ | | Wang et al. (2017) | Q | 80, 239 |
| LAMZNFDUTXWEFA-UHFFFAOYSA-N | $1.5\times10^7$ | | Wang et al. (2017) | Q | 80, 240 |
| MCM:DNOETOLOOH | $2.5\times10^{16}$ | | Wang et al. (2017) | Q | 80, 238 |
| $C_9H_{12}N_2O_{10}$ | $3.9\times10^{10}$ | | Wang et al. (2017) | Q | 80, 239 |
| KMVIJWGCTFHDMC-UHFFFAOYSA-N | $7.6\times10^5$ | | Wang et al. (2017) | Q | 80, 240 |
| MCM:DNPBNZLOOH | $2.9\times10^{16}$ | | Wang et al. (2017) | Q | 80, 238 |
| $C_9H_{12}N_2O_{10}$ | $1.9\times10^{10}$ | | Wang et al. (2017) | Q | 80, 239 |
| YOHJWFCUZUQBMA-UHFFFAOYSA-N | $2.8\times10^7$ | | Wang et al. (2017) | Q | 80, 240 |
| MCM:DNPETOLOOH | $2.1\times10^{16}$ | | Wang et al. (2017) | Q | 80, 238 |
| $C_9H_{12}N_2O_{10}$ | $1.9\times10^{10}$ | | Wang et al. (2017) | Q | 80, 239 |
| IOZSITWJVFMIBH-UHFFFAOYSA-N | $2.3\times10^4$ | | Wang et al. (2017) | Q | 80, 240 |
| MCM:DNT123LOOH | $1.7\times10^{16}$ | | Wang et al. (2017) | Q | 80, 238 |
| $C_9H_{12}N_2O_{10}$ | $3.9\times10^{10}$ | | Wang et al. (2017) | Q | 80, 239 |
| GBPJMJLXQYJDMC-UHFFFAOYSA-N | $4.5\times10^4$ | | Wang et al. (2017) | Q | 80, 240 |
| MCM:DNT124LOOH | $1.4\times10^{16}$ | | Wang et al. (2017) | Q | 80, 238 |
| $C_9H_{12}N_2O_{10}$ | $2.0\times10^{10}$ | | Wang et al. (2017) | Q | 80, 239 |
| INCIWBVFUFSSNB-UHFFFAOYSA-N | $1.6\times10^4$ | | Wang et al. (2017) | Q | 80, 240 |
| MCM:EMPONO2OOH | $1.4\times10^{14}$ | | Wang et al. (2017) | Q | 80, 238 |
| $C_9H_{13}NO_8$ | $1.5\times10^9$ | | Wang et al. (2017) | Q | 80, 239 |
| FZXCIHOWKZKJRK-UHFFFAOYSA-N | $6.3\times10^4$ | | Wang et al. (2017) | Q | 80, 240 |
| MCM:IPNCATCOOH | $2.5\times10^{17}$ | | Wang et al. (2017) | Q | 80, 238 |
| $C_9H_{13}NO_9$ | $8.3\times10^{10}$ | | Wang et al. (2017) | Q | 80, 239 |
| NYRKVILWQFQUNU-UHFFFAOYSA-N | $9.3\times10^6$ | | Wang et al. (2017) | Q | 80, 240 |
| MCM:MTNCATCOOH | $1.5\times10^{17}$ | | Wang et al. (2017) | Q | 80, 238 |
| $C_9H_{13}NO_9$ | $3.6\times10^{10}$ | | Wang et al. (2017) | Q | 80, 239 |
| MCLPDHQAEZEULP-UHFFFAOYSA-N | $1.4\times10^6$ | | Wang et al. (2017) | Q | 80, 240 |
| MCM:OTNCATCOOH | $3.6\times10^{17}$ | | Wang et al. (2017) | Q | 80, 238 |
| $C_9H_{13}NO_9$ | $7.1\times10^9$ | | Wang et al. (2017) | Q | 80, 239 |
| MVMRAYXEOTUUSG-UHFFFAOYSA-N | $2.5\times10^5$ | | Wang et al. (2017) | Q | 80, 240 |
| MCM:PNCATECOOH | $2.1\times10^{17}$ | | Wang et al. (2017) | Q | 80, 238 |
| $C_9H_{13}NO_9$ | $6.6\times10^{10}$ | | Wang et al. (2017) | Q | 80, 239 |
| AWPUGBPLGLTYEJ-UHFFFAOYSA-N | $5.9\times10^7$ | | Wang et al. (2017) | Q | 80, 240 |
| MCM:PTNCATCOOH | $2.8\times10^{17}$ | | Wang et al. (2017) | Q | 80, 238 |
| $C_9H_{13}NO_9$ | $3.5\times10^9$ | | Wang et al. (2017) | Q | 80, 239 |
| IJEUJFHQTRRKRP-UHFFFAOYSA-N | $9.6\times10^6$ | | Wang et al. (2017) | Q | 80, 240 |





Table A4.8: Nitro compounds ($RNO_2$) (...continued)

| Substance Formula (Trivial Name) [CAS Registry Number] InChIKey | $H_s^{cp}$ (at $T^{\ominus}$) $\left[\dfrac{\mathrm{mol}}{\mathrm{m^3\,Pa}}\right]$ | $\dfrac{\mathrm{d}\ln H_s^{cp}}{\mathrm{d}(1/T)}$ [K] | Reference | Type | Note |
|---|---|---|---|---|---|
| nitrobenzene | $4.7\times10^{-1}$ | 6100 | Brockbank (2013) | L | 1 |
| $C_6H_5NO_2$ | $6.7\times10^{-1}$ | | Chao et al. (2017) | M | |
| [98-95-3] | $6.4\times10^{-1}$ | 7500 | Hiatt (2013) | M | |
| LQNUZADURLCDLV-UHFFFAOYSA-N | $1.4\times10^{-1}$ | | Zhang et al. (2013) | M | 325 |
| | $2.3\times10^{-2}$ | 11000 | Dewulf et al. (1999) | M | 594 |
| | 1.2 | | Altschuh et al. (1999) | M | |
| | $1.4\times10^{-1}$ | | Hellmann (1987) | M | 87 |
| | $4.1\times10^{-1}$ | | Warner et al. (1980) | M | |
| | $4.7\times10^{-1}$ | | Chao et al. (2017) | V | |
| | $4.8\times10^{-1}$ | 6400 | Bernauer et al. (2006) | V | 1 |
| | $7.7\times10^{-1}$ | | Mackay et al. (2006d) | V | |
| | $4.2\times10^{-1}$ | | Lide and Frederikse (1995) | V | |
| | $7.7\times10^{-1}$ | | Mackay et al. (1995) | V | |
| | $4.6\times10^{-1}$ | | Hwang et al. (1992) | V | |
| | $7.8\times10^{-1}$ | | Yoshida et al. (1983) | V | |
| | $4.3\times10^{-1}$ | | Warner et al. (1980) | V | |
| | $4.2\times10^{-1}$ | | Hine and Mookerjee (1975) | V | |
| | $4.7\times10^{-1}$ | 4500 | Goldstein (1982) | X | 298 |
| | $4.2\times10^{-1}$ | | Hilal et al. (2008) | C | |
| | $4.1\times10^{-1}$ | | Schüürmann (2000) | C | 21 |
| | $7.5\times10^{-1}$ | | Mackay et al. (1995) | C | |
| | $7.5\times10^{-1}$ | | Ryan et al. (1988) | C | |
| | $4.1\times10^{-1}$ | | Shen (1982) | C | |
| | 1.1 | | Keshavarz et al. (2022) | Q | |
| | 2.1 | | Duchowicz et al. (2020) | Q | 299 |
| | $2.2\times10^{-1}$ | | Hilal et al. (2008) | Q | |
| | $2.7\times10^{-1}$ | | Modarresi et al. (2007) | Q | 67 |
| | | 4600 | Kühne et al. (2005) | Q | |
| | $4.2\times10^{-1}$ | | Yaffe et al. (2003) | Q | 248, 249 |
| | $5.1\times10^{-1}$ | | Yao et al. (2002) | Q | 229 |
| | $2.9\times10^{-1}$ | | Katritzky et al. (1998) | Q | |
| | 3.3 | | Nirmalakhandan et al. (1997) | Q | |
| | $2.0\times10^{-1}$ | | Russell et al. (1992) | Q | 279 |
| | $7.0\times10^{-1}$ | | Suzuki et al. (1992) | Q | 232 |
| | $4.1\times10^{-1}$ | | Duchowicz et al. (2020) | ? | 185, 21 |
| | | 5600 | Kühne et al. (2005) | ? | |
| | $4.7\times10^{-1}$ | | Yaws (1999) | ? | 21 |
| | $4.2\times10^{-1}$ | | Abraham et al. (1990) | ? | |
| nitrobenzene-d5 | $8.5\times10^{-1}$ | 7500 | Hiatt (2013) | M | |
| $C_6D_5NO_2$ | | | | | |
| [4165-60-0] | | | | | |
| LQNUZADURLCDLV-RALIUCGRSA-N | | | | | |



Table A4.8: Nitro compounds ($RNO_2$) (...continued)

| Substance Formula (Trivial Name) [CAS Registry Number] InChIKey | $H_s^{cp}$ (at $T^\ominus$) $\left[\dfrac{\text{mol}}{\text{m}^3\,\text{Pa}}\right]$ | $\dfrac{\text{d}\ln H_s^{cp}}{\text{d}(1/T)}$ [K] | Reference | Type | Note |
|---|---|---|---|---|---|
| 2-nitrotoluene | $2.4\times10^{-1}$ | 5800 | Brockbank (2013) | L | 1 |
| $C_6H_4(NO_2)CH_3$ | $9.6\times10^{-1}$ | | Chao et al. (2017) | M | |
| [88-72-2] | $7.9\times10^{-1}$ | | Altschuh et al. (1999) | M | |
| PLAZTCDQAHEYBI-UHFFFAOYSA-N | $2.7\times10^{-1}$ | | Mackay et al. (2006d) | V | |
| | $1.9\times10^{-1}$ | | Schüürmann (2000) | V | |
| | $1.8\times10^{-1}$ | | Lide and Frederikse (1995) | V | |
| | $2.7\times10^{-1}$ | | Mackay et al. (1995) | V | |
| | $1.7\times10^{-1}$ | | Hine and Mookerjee (1975) | V | |
| | $7.7\times10^{-2}$ | 2900 | Goldstein (1982) | X | 298 |
| | 1.5 | | Keshavarz et al. (2022) | Q | |
| | 1.2 | | Duchowicz et al. (2020) | Q | 299 |
| | $4.9\times10^{-1}$ | | Raventos-Duran et al. (2010) | Q | 242, 243 |
| | $2.5\times10^{-1}$ | | Raventos-Duran et al. (2010) | Q | 244 |
| | $3.9\times10^{-1}$ | | Raventos-Duran et al. (2010) | Q | 245 |
| | $4.2\times10^{-1}$ | | Zhang et al. (2010) | Q | 287, 288 |
| | $2.4\times10^{-1}$ | | Zhang et al. (2010) | Q | 287, 289 |
| | $2.5\times10^{-1}$ | | Zhang et al. (2010) | Q | 287, 290 |
| | $1.8\times10^{-1}$ | | Zhang et al. (2010) | Q | 287, 291 |
| | $4.2\times10^{-1}$ | | Zhang et al. (2010) | Q | 287, 288 |
| | $2.4\times10^{-1}$ | | Zhang et al. (2010) | Q | 287, 289 |
| | $2.2\times10^{-1}$ | | Zhang et al. (2010) | Q | 287, 290 |
| | $1.8\times10^{-1}$ | | Zhang et al. (2010) | Q | 287, 291 |
| | $1.4\times10^{-1}$ | | Hilal et al. (2008) | Q | |
| | $2.3\times10^{-1}$ | | Modarresi et al. (2007) | Q | 67 |
| | | 4900 | Kühne et al. (2005) | Q | |
| | $1.4\times10^{-1}$ | | Yaffe et al. (2003) | Q | 248, 272 |
| | 2.3 | | Nirmalakhandan et al. (1997) | Q | |
| | $5.7\times10^{-1}$ | | Suzuki et al. (1992) | Q | 232 |
| | $7.9\times10^{-1}$ | | Duchowicz et al. (2020) | ? | 185, 21 |
| | | 5900 | Kühne et al. (2005) | ? | |
| | $1.7\times10^{-1}$ | | Abraham et al. (1990) | ? | |
| 3-nitrotoluene | $2.7\times10^{-1}$ | 7000 | Brockbank (2013) | L | 1 |
| $C_6H_4(NO_2)CH_3$ | 1.3 | | Chao et al. (2017) | M | |
| [99-08-1] | 1.1 | | Altschuh et al. (1999) | M | |
| QZYHIOPPLUPUJF-UHFFFAOYSA-N | $2.8\times10^{-1}$ | | Li and Carr (1993) | M | |
| | $1.3\times10^{-1}$ | | Mackay et al. (2006d) | V | |
| | $1.3\times10^{-1}$ | | Mackay et al. (1995) | V | |
| | $1.4\times10^{-1}$ | | Hine and Mookerjee (1975) | V | |
| | $1.4\times10^{-1}$ | 3200 | Goldstein (1982) | X | 298 |
| | 1.5 | | Keshavarz et al. (2022) | Q | |
| | 1.2 | | Duchowicz et al. (2020) | Q | 299 |
| | $4.2\times10^{-1}$ | | Zhang et al. (2010) | Q | 287, 288 |
| | $2.5\times10^{-1}$ | | Zhang et al. (2010) | Q | 287, 289 |
| | $4.1\times10^{-1}$ | | Zhang et al. (2010) | Q | 287, 290 |
| | $1.8\times10^{-1}$ | | Zhang et al. (2010) | Q | 287, 291 |
| | $1.8\times10^{-1}$ | | Hilal et al. (2008) | Q | |
| | $1.6\times10^{-1}$ | | Modarresi et al. (2007) | Q | 67 |



Table A4.8: Nitro compounds ($RNO_2$) (...continued)

| Substance Formula (Trivial Name) [CAS Registry Number] InChIKey | $H_s^{cp}$ (at $T^\ominus$) $\left[\dfrac{mol}{m^3\,Pa}\right]$ | $\dfrac{d\ln H_s^{cp}}{d(1/T)}$ [K] | Reference | Type | Note |
|---|---|---|---|---|---|
| | | 4900 | Kühne et al. (2005) | Q | |
| | $1.4\times10^{-1}$ | | Yaffe et al. (2003) | Q | 248, 249 |
| | 2.3 | | Nirmalakhandan et al. (1997) | Q | |
| | $5.7\times10^{-1}$ | | Suzuki et al. (1992) | Q | 232 |
| | 1.1 | | Duchowicz et al. (2020) | ? | 185, 21 |
| | | 4900 | Kühne et al. (2005) | ? | |
| | $1.3\times10^{-1}$ | | Yaws (1999) | ? | 21, 38 |
| | $1.4\times10^{-1}$ | | Abraham et al. (1990) | ? | |
| 4-nitrotoluene $C_6H_4(NO_2)CH_3$ [99-99-0] ZPTVNYMJQHSSEA-UHFFFAOYSA-N | $8.9\times10^{-1}$ | 7100 | Brockbank (2013) | L | 1 |
| | 1.8 | | Altschuh et al. (1999) | M | |
| | 2.8 | | Mackay et al. (2006d) | V | |
| | $2.0\times10^{-1}$ | | Lide and Frederikse (1995) | V | |
| | 2.8 | | Mackay et al. (1995) | V | |
| | $1.6\times10^{-1}$ | 3100 | Goldstein (1982) | X | 298 |
| | 1.5 | | Keshavarz et al. (2022) | Q | |
| | 1.2 | | Duchowicz et al. (2020) | Q | 184 |
| | $1.4\times10^{-1}$ | | Li et al. (2014) | Q | 241 |
| | $4.2\times10^{-1}$ | | Zhang et al. (2010) | Q | 287, 288 |
| | $2.8\times10^{-1}$ | | Zhang et al. (2010) | Q | 287, 289 |
| | $9.0\times10^{-1}$ | | Zhang et al. (2010) | Q | 287, 290 |
| | $1.8\times10^{-1}$ | | Zhang et al. (2010) | Q | 287, 291 |
| | $2.0\times10^{-1}$ | | Hilal et al. (2008) | Q | |
| | $2.4\times10^{-1}$ | | Modarresi et al. (2007) | Q | 67 |
| | | 4900 | Kühne et al. (2005) | Q | |
| | 1.8 | | Duchowicz et al. (2020) | ? | 185, 21 |
| | | 3800 | Kühne et al. (2005) | ? | |
| 1,2-dinitrobenzene $C_6H_4N_2O_4$ [528-29-0] IZUKQUVSCNEFMJ-UHFFFAOYSA-N | $1.9\times10^2$ | | Duchowicz et al. (2020) | V | 186 |
| | $1.9\times10^2$ | | HSDB (2015) | V | |
| | $3.5\times10^2$ | | Duchowicz et al. (2020) | Q | |
| | $1.2\times10^2$ | | Zhang et al. (2010) | Q | 287, 288 |
| | $3.2\times10^1$ | | Zhang et al. (2010) | Q | 287, 289 |
| | $2.6\times10^1$ | | Zhang et al. (2010) | Q | 287, 290 |
| | $2.7\times10^1$ | | Zhang et al. (2010) | Q | 287, 291 |
| 1,3-dinitrobenzene $C_6H_4N_2O_4$ [99-65-0] WDCYWAQPCXBPJA-UHFFFAOYSA-N | $1.8\times10^2$ | | Chao et al. (2017) | M | |
| | $2.0\times10^2$ | | Altschuh et al. (1999) | M | |
| | | | Mackay et al. (2006d) | V | 558 |
| | $5.0\times10^2$ | | Mackay et al. (1995) | V | |
| | $3.9\times10^1$ | | Smith et al. (1981a) | V | |
| | $2.5\times10^2$ | | Keshavarz et al. (2022) | Q | |
| | $2.1\times10^2$ | | Duchowicz et al. (2020) | Q | |
| | $2.0\times10^2$ | | Gharagheizi et al. (2012) | Q | |
| | $1.2\times10^2$ | | Raventos-Duran et al. (2010) | Q | 242, 243 |
| | $1.6\times10^1$ | | Raventos-Duran et al. (2010) | Q | 244 |
| | $1.2\times10^2$ | | Raventos-Duran et al. (2010) | Q | 245 |
| | $2.0\times10^2$ | | Duchowicz et al. (2020) | ? | 185, 21 |



Table A4.8: Nitro compounds ($RNO_2$) (...continued)

| Substance Formula (Trivial Name) [CAS Registry Number] InChIKey | $H_s^{cp}$ (at $T^{\ominus}$) $\left[\dfrac{\mathrm{mol}}{\mathrm{m}^3\,\mathrm{Pa}}\right]$ | $\dfrac{\mathrm{d}\ln H_s^{cp}}{\mathrm{d}(1/T)}$ [K] | Reference | Type | Note |
|---|---|---|---|---|---|
| 1,4-dinitrobenzene $C_6H_4N_2O_4$ [100-25-4] FYFDQJRXFWGIBS-UHFFFAOYSA-N | $2.0\times10^{-1}$ $2.0\times10^{-1}$ $1.2\times10^{2}$ | | Mackay et al. (2006d) Mackay et al. (1995) HSDB (2015) | V V Q | 99 |
| 1,3,5-trinitrobenzene $C_6H_3N_3O_6$ [99-35-4] UATJOMSPNYCXIX-UHFFFAOYSA-N | $1.5\times10^{3}$ $2.7\times10^{2}$ $2.8\times10^{2}$ | | HSDB (2015) Yaws (2003) Gharagheizi et al. (2010) | V X Q | 237, 80 246 |
| 2,4,6-trinitrophenol $C_6H_3N_3O_7$ (picric acid) [88-89-1] OXNIZHLAWKMVMX-UHFFFAOYSA-N | $3.0\times10^{5}$ | | Ebert et al. (2023) | ? | 316 |
| 2-nitrobenzenamine $C_6H_6N_2O_2$ (2-nitroaniline) [88-74-4] DPJCXCZTLWNFOH-UHFFFAOYSA-N | $7.1\times10^{1}$ $1.7\times10^{2}$ $1.0\times10^{2}$ $2.0\times10^{2}$ $2.5\times10^{2}$ $3.1\times10^{1}$ $2.1\times10^{2}$ $4.5\times10^{2}$ $1.7\times10^{2}$ | 6800 | Brockbank (2013) Altschuh et al. (1999) Abraham et al. (1994a) Keshavarz et al. (2022) Duchowicz et al. (2020) Hilal et al. (2008) Modarresi et al. (2007) Nirmalakhandan et al. (1997) Duchowicz et al. (2020) | L M R Q Q Q Q Q ? | 1 184 67 185, 21 |
| 3-nitrobenzenamine $C_6H_6N_2O_2$ (3-nitroaniline) [99-09-2] XJCVRTZCHMZPBD-UHFFFAOYSA-N | $6.9\times10^{2}$ $1.2\times10^{3}$ $4.2\times10^{3}$ $1.5\times10^{3}$ $2.7\times10^{3}$ $4.0\times10^{2}$ $6.0\times10^{2}$ $4.4\times10^{2}$ $1.3\times10^{3}$ $1.2\times10^{3}$ $1.2\times10^{3}$ | | Meylan and Howard (1991) Abraham et al. (1994a) Keshavarz et al. (2022) Duchowicz et al. (2020) Hilal et al. (2008) Modarresi et al. (2007) English and Carroll (2001) Nirmalakhandan et al. (1997) Meylan and Howard (1991) Duchowicz et al. (2020) HSDB (2015) | V R Q Q Q Q Q Q Q ? ? | 67 230, 231 185, 21 419 |
| 4-nitrobenzenamine $C_6H_6N_2O_2$ (4-nitroaniline) [100-01-6] TYMLOMAKGOJONV-UHFFFAOYSA-N | $8.7\times10^{3}$ $8.6\times10^{3}$ $1.4\times10^{4}$ $1.8\times10^{2}$ $4.2\times10^{3}$ $1.5\times10^{3}$ $8.5\times10^{2}$ $1.7\times10^{2}$ $2.2\times10^{3}$ $1.5\times10^{2}$ $4.4\times10^{2}$ $7.8\times10^{3}$ | 8200 | Brockbank (2013) Altschuh et al. (1999) Abraham et al. (1994a) Yaws (2003) Keshavarz et al. (2022) Duchowicz et al. (2020) Gharagheizi et al. (2012) Gharagheizi et al. (2010) Hilal et al. (2008) Modarresi et al. (2007) Nirmalakhandan et al. (1997) Duchowicz et al. (2020) | L M R X Q Q Q Q Q Q Q ? | 1 237 246 67 185, 21 |





Table A4.8: Nitro compounds ($RNO_2$) (...continued)

| Substance Formula (Trivial Name) [CAS Registry Number] InChIKey | $H_s^{cp}$ (at $T^{\ominus}$) $\left[\dfrac{\mathrm{mol}}{\mathrm{m}^3\,\mathrm{Pa}}\right]$ | $\dfrac{\mathrm{d}\ln H_s^{cp}}{\mathrm{d}(1/T)}$ [K] | Reference | Type | Note |
|---|---|---|---|---|---|
| 2,4-dinitrobenzenamine $C_6H_5N_3O_4$ [97-02-9] LXQOQPGNCGEELI-UHFFFAOYSA-N | $6.5\times10^4$ | | HSDB (2015) | Q | 545 |
| 2-methyl-6-nitroaniline $C_7H_8N_2O_2$ [570-24-1] FCMRHMPITHLLLA-UHFFFAOYSA-N | $4.6\times10^1$ | | Abraham et al. (2019) | Q | |
| 1-methyl-2,3-dinitrobenzene $C_7H_6N_2O_4$ (2,3-dinitrotoluene; 2,3-DNT) [602-01-7] DYSXLQBUUOPLBB-UHFFFAOYSA-N | $1.1\times10^2$ | | HSDB (2015) | Q | 447 |
| | $1.1\times10^2$ | | Zhang et al. (2010) | Q | 287, 288 |
| | $2.2\times10^1$ | | Zhang et al. (2010) | Q | 287, 289 |
| | 9.5 | | Zhang et al. (2010) | Q | 287, 290 |
| | $1.5\times10^1$ | | Zhang et al. (2010) | Q | 287, 291 |
| | $1.1\times10^2$ | | Zhang et al. (2010) | Q | 287, 288 |
| | $2.3\times10^1$ | | Zhang et al. (2010) | Q | 287, 289 |
| | $1.1\times10^1$ | | Zhang et al. (2010) | Q | 287, 290 |
| | $1.5\times10^1$ | | Zhang et al. (2010) | Q | 287, 291 |
| 1-methyl-2,4-dinitrobenzene $C_7H_6N_2O_4$ (2,4-dinitrotoluene; 2,4-DNT) [121-14-2] RMBFBMJGBANMMK-UHFFFAOYSA-N | $4.0\times10^1$ | 7900 | Brockbank (2013) | L | 1 |
| | $1.8\times10^2$ | | Altschuh et al. (1999) | M | |
| | $1.1\times10^1$ | | Mackay et al. (2006d) | V | |
| | $1.0\times10^2$ | | Schüürmann (2000) | V | |
| | $1.1\times10^1$ | | Mackay et al. (1995) | V | |
| | $6.3\times10^1$ | | Smith et al. (1981a) | V | |
| | $2.1\times10^{-1}$ | 2900 | Goldstein (1982) | X | 298 |
| | 2.2 | | Mackay et al. (1995) | C | |
| | $3.1\times10^{-2}$ | | Ryan et al. (1988) | C | |
| | $3.4\times10^2$ | | Keshavarz et al. (2022) | Q | |
| | $1.6\times10^2$ | | Duchowicz et al. (2020) | Q | |
| | $9.9\times10^1$ | | Raventos-Duran et al. (2010) | Q | 242, 243 |
| | $1.6\times10^1$ | | Raventos-Duran et al. (2010) | Q | 244 |
| | $9.9\times10^1$ | | Raventos-Duran et al. (2010) | Q | 245 |
| | $1.1\times10^2$ | | Zhang et al. (2010) | Q | 287, 288 |
| | $1.6\times10^1$ | | Zhang et al. (2010) | Q | 287, 289 |
| | 5.0 | | Zhang et al. (2010) | Q | 287, 290 |
| | $1.5\times10^1$ | | Zhang et al. (2010) | Q | 287, 291 |
| | $1.8\times10^2$ | | Duchowicz et al. (2020) | ? | 185, 21 |
| 2-methyl-1,4-dinitrobenzene $C_7H_6N_2O_4$ (2,5-dinitrotoluene; 2,5-DNT) [619-15-8] KZBOXYKTSUUBTO-UHFFFAOYSA-N | $1.8\times10^1$ | | HSDB (2015) | V | |
| | $1.1\times10^2$ | | Zhang et al. (2010) | Q | 287, 288 |
| | $1.8\times10^1$ | | Zhang et al. (2010) | Q | 287, 289 |
| | 1.4 | | Zhang et al. (2010) | Q | 287, 290 |
| | $1.5\times10^1$ | | Zhang et al. (2010) | Q | 287, 291 |



Table A4.8: Nitro compounds ($RNO_2$) (... continued)

| Substance Formula (Trivial Name) [CAS Registry Number] InChIKey | $H_s^{cp}$ (at $T^\ominus$) $\left[\dfrac{\text{mol}}{\text{m}^3\,\text{Pa}}\right]$ | $\dfrac{\text{d}\ln H_s^{cp}}{\text{d}(1/T)}$ [K] | Reference | Type | Note |
|---|---|---|---|---|---|
| 2-methyl-1,3-dinitrobenzene | 4.8 | 7600 | Brockbank (2013) | L | 1 |
| $C_7H_6N_2O_4$ | $1.3\times10^1$ | | Duchowicz et al. (2020) | V | 186 |
| (2,6-dinitrotoluene; 2,6-DNT) | $1.5\times10^1$ | | HSDB (2015) | V | |
| [606-20-2] | $1.4\times10^1$ | | Mackay et al. (2006d) | V | |
| XTRDKALNCIHHNI-UHFFFAOYSA-N | $1.4\times10^1$ | | Mackay et al. (1995) | V | |
| | 1.2 | | Mackay et al. (1995) | C | |
| | $3.1\times10^{-2}$ | | Ryan et al. (1988) | C | |
| | $1.6\times10^2$ | | Duchowicz et al. (2020) | Q | |
| | 1.6 | | Li et al. (2014) | Q | 241 |
| | $1.1\times10^2$ | | Zhang et al. (2010) | Q | 287, 288 |
| | $2.1\times10^1$ | | Zhang et al. (2010) | Q | 287, 289 |
| | 4.3 | | Zhang et al. (2010) | Q | 287, 290 |
| | $1.5\times10^1$ | | Zhang et al. (2010) | Q | 287, 291 |
| 4-methyl-1,2-dinitrobenzene | $3.3\times10^1$ | 9600 | Brockbank (2013) | L | 1 |
| $C_7H_6N_2O_4$ | $1.1\times10^2$ | | HSDB (2015) | Q | 447 |
| (3,4-dinitrotoluene; 3,4-DNT) | $1.1\times10^2$ | | Zhang et al. (2010) | Q | 287, 288 |
| [610-39-9] | $3.9\times10^1$ | | Zhang et al. (2010) | Q | 287, 289 |
| INYDMNPNDHRJQJ-UHFFFAOYSA-N | $3.1\times10^1$ | | Zhang et al. (2010) | Q | 287, 290 |
| | $1.5\times10^1$ | | Zhang et al. (2010) | Q | 287, 291 |
| 1-methyl-2,4,6-trinitrobenzene | $4.7\times10^2$ | 7700 | Brockbank (2013) | L | 1 |
| $C_7H_5N_3O_6$ | $4.7\times10^2$ | | HSDB (2015) | V | |
| (2,4,6-trinitrotoluene; TNT) | $5.4\times10^2$ | | Schüürmann (2000) | V | |
| [118-96-7] | | 6200 | Kühne et al. (2005) | Q | |
| SPSSULHKWOKEEL-UHFFFAOYSA-N | | 6400 | Kühne et al. (2005) | ? | |
| 2,4,6-trinitro-1,3-dimethyl-5-*tert*-butylbenzene | $3.2\times10^{-1}$ | | Lee et al. (2012) | M | |
| $C_{12}H_{15}N_3O_6$ | $1.7\times10^{-2}$ | | Amoore and Buttery (1978) | V | |
| (musk xylene) | $1.3\times10^3$ | | HSDB (2015) | Q | 99 |
| [81-15-2] | $9.5\times10^3$ | | Zhang et al. (2010) | Q | 287, 288 |
| XMWRWTSZNLOZFN-UHFFFAOYSA-N | 5.6 | | Zhang et al. (2010) | Q | 287, 289 |
| | $4.8\times10^{-2}$ | | Zhang et al. (2010) | Q | 287, 290 |
| | $1.5\times10^2$ | | Zhang et al. (2010) | Q | 287, 291 |
| 2-nitrophenol | 9.9 | | Chao et al. (2017) | M | |
| $HOC_6H_4(NO_2)$ | 1.4 | 5700 | Guo and Brimblecombe (2007) | M | |
| [88-75-5] | $8.3\times10^{-1}$ | 6300 | Harrison et al. (2002) | M | |
| IQUPABOKLQSFBK-UHFFFAOYSA-N | $8.9\times10^{-1}$ | 6300 | Müller and Heal (2001) | M | |
| | $7.7\times10^{-1}$ | | Tremp et al. (1993) | M | 12 |
| | $6.1\times10^{-1}$ | | Mackay et al. (2006c) | V | |
| | 2.9 | | Lide and Frederikse (1995) | V | |
| | $7.9\times10^{-1}$ | | Riederer (1990) | V | |
| | $7.3\times10^{-1}$ | | Schwarzenbach et al. (1988) | V | 12 |
| | 2.8 | | Leuenberger et al. (1985) | V | 416 |
| | $9.2\times10^{-1}$ | | Abraham et al. (1994a) | R | |
| | $6.9\times10^{-1}$ | 4600 | Goldstein (1982) | X | 298 |
| | 1.3 | | Ryan et al. (1988) | C | |





Table A4.8: Nitro compounds ($RNO_2$) (...continued)

| Substance Formula (Trivial Name) [CAS Registry Number] InChIKey | $H_s^{cp}$ (at $T^\ominus$) $\left[\dfrac{\text{mol}}{\text{m}^3\,\text{Pa}}\right]$ | $\dfrac{\text{d}\ln H_s^{cp}}{\text{d}(1/T)}$ [K] | Reference | Type | Note |
|---|---|---|---|---|---|
| | 1.1 | | Abraham et al. (2019) | Q | |
| | 6.6 | | Wang et al. (2017) | Q | 80, 238 |
| | 2.0 | | Wang et al. (2017) | Q | 80, 239 |
| | $2.1\times10^{-1}$ | | Wang et al. (2017) | Q | 80, 240 |
| | $3.9\times10^{-1}$ | | Raventos-Duran et al. (2010) | Q | 242, 243 |
| | $7.8\times10^{-1}$ | | Raventos-Duran et al. (2010) | Q | 244 |
| | 1.6 | | Raventos-Duran et al. (2010) | Q | 245 |
| | 5.3 | | Hilal et al. (2008) | Q | |
| | | 4400 | Kühne et al. (2005) | Q | |
| | $3.5\times10^1$ | | Katritzky et al. (1998) | Q | |
| | $1.5\times10^4$ | | Nirmalakhandan et al. (1997) | Q | |
| | | 6300 | Kühne et al. (2005) | ? | |
| | $7.0\times10^{-1}$ | | Abraham et al. (1990) | ? | |
| 3-nitrophenol $HOC_6H_4(NO_2)$ [554-84-7] RTZZCYNQPHTPPL-UHFFFAOYSA-N | $1.6\times10^2$ | | Guo and Brimblecombe (2007) | M | 555 |
| | 1.0 | | Lide and Frederikse (1995) | V | |
| | $4.9\times10^3$ | | Gaffney and Senum (1984) | X | 389 |
| | $2.1\times10^4$ | | Keshavarz et al. (2022) | Q | |
| | $1.4\times10^4$ | | Duchowicz et al. (2020) | Q | 299 |
| | $2.2\times10^3$ | | Abraham et al. (2019) | Q | |
| | $2.0\times10^3$ | | Raventos-Duran et al. (2010) | Q | 242, 243 |
| | $1.2\times10^4$ | | Raventos-Duran et al. (2010) | Q | 244 |
| | $4.9\times10^3$ | | Raventos-Duran et al. (2010) | Q | 245 |
| | $9.5\times10^3$ | | Hilal et al. (2008) | Q | |
| | $2.1\times10^2$ | | Modarresi et al. (2007) | Q | 67 |
| | $4.8\times10^3$ | | English and Carroll (2001) | Q | 230, 231 |
| | $1.5\times10^4$ | | Nirmalakhandan et al. (1997) | Q | |
| | $4.9\times10^3$ | | Duchowicz et al. (2020) | ? | 185, 21 |
| | $4.6\times10^3$ | | Abraham et al. (1990) | ? | |
| 4-nitrophenol $HOC_6H_4(NO_2)$ [100-02-7] BTJIUGUIPKRLHP-UHFFFAOYSA-N | $1.4\times10^1$ | | Chao et al. (2017) | M | |
| | $2.1\times10^2$ | | Guo and Brimblecombe (2007) | M | 555 |
| | $7.7\times10^2$ | | Tremp et al. (1993) | M | 12 |
| | $3.0\times10^2$ | | Lide and Frederikse (1995) | V | |
| | $2.0\times10^4$ | | Riederer (1990) | V | |
| | $3.0\times10^2$ | | Schwarzenbach et al. (1988) | V | 12 |
| | $9.4\times10^4$ | | Yoshida et al. (1983) | V | |
| | $2.6\times10^4$ | 9100 | Parsons et al. (1971) | T | 417 |
| | 9.8 | 6000 | Goldstein (1982) | X | 298 |
| | 1.6 | | Ryan et al. (1988) | C | |
| | $2.1\times10^4$ | | Keshavarz et al. (2022) | Q | |
| | $2.2\times10^4$ | | Duchowicz et al. (2020) | Q | 184 |
| | $1.4\times10^4$ | | Abraham et al. (2019) | Q | |
| | $2.4\times10^4$ | | Li et al. (2014) | Q | 241 |
| | $2.0\times10^3$ | | Raventos-Duran et al. (2010) | Q | 242, 243 |
| | $3.9\times10^3$ | | Raventos-Duran et al. (2010) | Q | 244 |
| | $4.9\times10^3$ | | Raventos-Duran et al. (2010) | Q | 245 |
| | $6.1\times10^3$ | | Hilal et al. (2008) | Q | |





Table A4.8: Nitro compounds ($RNO_2$) (. . . continued)

| Substance<br>Formula<br>(Trivial Name)<br>[CAS Registry Number]<br>InChIKey | $H_s^{cp}$<br>(at $T^\ominus$)<br>$\left[\dfrac{mol}{m^3\,Pa}\right]$ | $\dfrac{d \ln H_s^{cp}}{d(1/T)}$<br><br>[K] | Reference | Type | Note |
|---|---|---|---|---|---|
| | $2.3\times10^2$ | | Modarresi et al. (2007) | Q | 67 |
| | $1.5\times10^4$ | | Nirmalakhandan et al. (1997) | Q | |
| | $2.4\times10^4$ | | Duchowicz et al. (2020) | ? | 185, 21 |
| | $2.6\times10^4$ | | Abraham et al. (1990) | ? | |
| 4-nitroanisole<br>$C_7H_7NO_3$<br>[100-17-4]<br>BNUHAJGCKIQFGE-UHFFFAOYSA-N | 5.0 | | Ebert et al. (2023) | ? | 365 |
| 3-methyl-2-nitrophenol<br>$C_7H_7NO_3$<br>[4920-77-8]<br>QIORDSKCCHRSSD-UHFFFAOYSA-N | 3.2<br>2.4<br>$1.9\times10^2$ | <br><br><br>4700<br>4200 | Tremp et al. (1993)<br>Schwarzenbach et al. (1988)<br>Modarresi et al. (2007)<br>Kühne et al. (2005)<br>Kühne et al. (2005) | M<br>V<br>Q<br>Q<br>? | 12<br>12<br>67<br><br> |
| 4-methyl-2-nitrophenol<br>$C_7H_7NO_3$<br>[119-33-5]<br>SYDNSSSQVSOXTN-UHFFFAOYSA-N | $6.7\times10^{-1}$<br>$6.1\times10^{-1}$<br>3.9<br>1.9<br>$2.5\times10^{-1}$<br>$3.1\times10^{-1}$<br>$6.2\times10^{-1}$<br>1.2 | <br><br><br><br><br><br><br><br>4700<br>6800 | Tremp et al. (1993)<br>Schwarzenbach et al. (1988)<br>Wang et al. (2017)<br>Wang et al. (2017)<br>Wang et al. (2017)<br>Raventos-Duran et al. (2010)<br>Raventos-Duran et al. (2010)<br>Raventos-Duran et al. (2010)<br>Kühne et al. (2005)<br>Kühne et al. (2005) | M<br>V<br>Q<br>Q<br>Q<br>Q<br>Q<br>Q<br>Q<br>? | 12<br>12<br>80, 238<br>80, 239<br>80, 240<br>242, 243<br>244<br>245<br><br> |
| 5-methyl-2-nitrophenol<br>$C_7H_7NO_3$<br>[700-38-9]<br>NQXUSSVLFOBRSE-UHFFFAOYSA-N | $7.7\times10^{-1}$<br>$6.7\times10^{-1}$<br>$3.1\times10^{-1}$<br>$7.8\times10^{-1}$<br>1.2 | <br><br><br><br><br>4700<br>5600 | Tremp et al. (1993)<br>Schwarzenbach et al. (1988)<br>Raventos-Duran et al. (2010)<br>Raventos-Duran et al. (2010)<br>Raventos-Duran et al. (2010)<br>Kühne et al. (2005)<br>Kühne et al. (2005) | M<br>V<br>Q<br>Q<br>Q<br>Q<br>? | 12<br>12<br>271, 243<br>244<br>245<br><br> |
| 6-methyl-2-nitrophenol<br>$C_7H_7NO_3$<br>[13073-29-5]<br>AQDKZPFDOWHRDZ-UHFFFAOYSA-N | $2.9\times10^{-1}$<br>3.9<br>$2.0\times10^1$<br>$1.2\times10^{-1}$<br>$3.1\times10^{-1}$<br>3.9<br>1.2 | <br><br><br><br><br><br><br>4700<br>5200 | Tremp et al. (1993)<br>Wang et al. (2017)<br>Wang et al. (2017)<br>Wang et al. (2017)<br>Raventos-Duran et al. (2010)<br>Raventos-Duran et al. (2010)<br>Raventos-Duran et al. (2010)<br>Kühne et al. (2005)<br>Kühne et al. (2005) | M<br>Q<br>Q<br>Q<br>Q<br>Q<br>Q<br>Q<br>? | 12<br>80, 238<br>80, 239<br>80, 240<br>242, 243<br>244<br>245<br><br> |
| 3-methyl-4-nitrophenol<br>$C_7H_7NO_3$<br>[2581-34-2]<br>PIIZYNQECPTVEO-UHFFFAOYSA-N | $6.2\times10^2$<br>$1.6\times10^3$<br>$3.9\times10^3$<br>$3.9\times10^3$<br>$2.1\times10^2$ | | Tremp et al. (1993)<br>Raventos-Duran et al. (2010)<br>Raventos-Duran et al. (2010)<br>Raventos-Duran et al. (2010)<br>Modarresi et al. (2007) | M<br>Q<br>Q<br>Q<br>Q | 12<br>271, 243<br>244<br>245<br>67 |



Table A4.8: Nitro compounds ($RNO_2$) (... continued)

| Substance Formula (Trivial Name) [CAS Registry Number] InChIKey | $H_s^{cp}$ (at $T^{\ominus}$) $\left[\dfrac{\mathrm{mol}}{\mathrm{m^3\,Pa}}\right]$ | $\dfrac{\mathrm{d}\ln H_s^{cp}}{\mathrm{d}(1/T)}$ [K] | Reference | Type | Note |
|---|---|---|---|---|---|
| 4-methoxy-2-nitrophenol $C_7H_7NO_4$ [1568-70-3] YBUGOACXDPDUIR-UHFFFAOYSA-N | 5.3 $2.3\times10^{-1}$ 6.2 $1.2\times10^2$ $2.5\times10^1$ $9.4\times10^1$ | 4900 6600 | Tremp et al. (1993) Schwarzenbach et al. (1988) Raventos-Duran et al. (2010) Raventos-Duran et al. (2010) Raventos-Duran et al. (2010) Modarresi et al. (2007) Kühne et al. (2005) Kühne et al. (2005) | M V Q Q Q Q Q ? | 12 12 242, 243 244 245 67 |
| 4-amino-2,6-dinitrotoluene $C_7H_7N_3O_4$ [19406-51-0] KQRJATLINVYHEZ-UHFFFAOYSA-N | $7.3\times10^3$ | | Ebert et al. (2023) | ? | 365 |
| 4-hydroxy-3-nitro-benzaldehyde $C_7H_5NO_4$ [3011-34-5] YTHJCZRFJGXPTL-UHFFFAOYSA-N | 9.4 $3.9\times10^1$ $2.0\times10^2$ $6.2\times10^2$ | | Schwarzenbach et al. (1988) Raventos-Duran et al. (2010) Raventos-Duran et al. (2010) Raventos-Duran et al. (2010) | V Q Q Q | 12 242, 243 244 245 |
| 2,4-dinitrophenol $C_6H_4N_2O_5$ [51-28-5] UFBJCMHMOXMLKC-UHFFFAOYSA-N | $9.7\times10^2$ $1.1\times10^2$ $3.5\times10^1$ $1.5\times10^4$ $2.0\times10^2$ $2.6\times10^3$ $2.1\times10^3$ 3.0 $7.8\times10^1$ $6.2\times10^2$ $3.9\times10^2$ $3.6\times10^2$ $6.2\times10^2$ 4.7 $1.3\times10^3$ $4.7\times10^1$ | 5000 3300 | Chao et al. (2017) Tremp et al. (1993) Schwarzenbach et al. (1988) Ryan et al. (1988) Abraham et al. (2019) Wang et al. (2017) Wang et al. (2017) Wang et al. (2017) Raventos-Duran et al. (2010) Raventos-Duran et al. (2010) Raventos-Duran et al. (2010) Zhang et al. (2010) Zhang et al. (2010) Zhang et al. (2010) Zhang et al. (2010) Modarresi et al. (2007) Kühne et al. (2005) Kühne et al. (2005) | M M V C Q Q Q Q Q Q Q Q Q Q Q Q Q ? | 12 12 80, 238 80, 239 80, 240 242, 243 244 245 287, 288 287, 289 287, 290 287, 291 67 |
| 2,5-dinitrophenol $C_6H_4N_2O_5$ [329-71-5] UWEZBKLLMKVIPI-UHFFFAOYSA-N | $1.5\times10^1$ $1.1\times10^2$ $7.8\times10^1$ $4.9\times10^2$ $3.9\times10^2$ | | Schwarzenbach et al. (1988) Abraham et al. (2019) Raventos-Duran et al. (2010) Raventos-Duran et al. (2010) Raventos-Duran et al. (2010) | V Q Q Q Q | 12 271, 243 244 245 |
| picramic acid $C_6H_5N_3O_5$ (4,6-dinitro-2-aminophenol) [96-91-3] QXYMVUZOGFVPGH-UHFFFAOYSA-N | $1.0\times10^6$ | | HSDB (2015) | Q | 99 |





Table A4.8: Nitro compounds ($RNO_2$) (...continued)

| Substance<br>Formula<br>(Trivial Name)<br>[CAS Registry Number]<br>InChIKey | $H_s^{cp}$<br>(at $T^{\ominus}$)<br>$\left[\dfrac{\text{mol}}{\text{m}^3\,\text{Pa}}\right]$ | $\dfrac{\mathrm{d}\ln H_s^{cp}}{\mathrm{d}(1/T)}$<br><br>[K] | Reference | Type | Note |
|---|---|---|---|---|---|
| 4-amino-2-nitrophenol<br>$C_6H_6N_2O_3$<br>[119-34-6]<br>WHODQVWERNSQEO-UHFFFAOYSA-N | $4.5\times10^6$ | | HSDB (2015) | Q | 99 |
| 2-amino-5-nitrophenol<br>$C_6H_6N_2O_3$<br>[121-88-0]<br>DOPJTDJKZNWLRB-UHFFFAOYSA-N | $1.3\times10^7$ | | HSDB (2015) | Q | 99 |
| 2-amino-4-nitrophenol<br>$C_6H_6N_2O_3$<br>[99-57-0]<br>VLZVIIYRNMWPSN-UHFFFAOYSA-N | $4.5\times10^6$ | | HSDB (2015) | Q | 99 |
| 4-nitro-$o$-phenylenediamine<br>$C_6H_7N_3O_2$<br>(4-nitro-1,2-diaminobenzene)<br>[99-56-9]<br>RAUWPNXIALNKQM-UHFFFAOYSA-N | $1.3\times10^6$ | | HSDB (2015) | Q | 99 |
| 4-nitrobenzene-1,3-diamine<br>$C_6H_7N_3O_2$<br>[5131-58-8]<br>DPIZKMGPXNXSGL-UHFFFAOYSA-N | $1.7\times10^5$ | | HSDB (2015) | Q | 99 |
| 2-nitro-1,4-benzenediamine<br>$C_6H_7N_3O_2$<br>[5307-14-2]<br>HVHNMNGARPCGGD-UHFFFAOYSA-N | $1.7\times10^5$ | | HSDB (2015) | Q | 99 |
| 4-methyl-2,6-dinitrophenol<br>$C_7H_6N_2O_5$<br>(2,6-dinitro-$p$-cresol)<br>[609-93-8]<br>HOYRZHJJAHRMLL-UHFFFAOYSA-N | $1.9\times10^2$<br>$3.2\times10^2$<br>$3.4\times10^3$<br>$8.8\times10^1$<br>8.0<br>$4.0\times10^1$<br><br> | <br><br><br><br><br><br>3000<br>3400 | Tremp et al. (1993)<br>Zhang et al. (2010)<br>Zhang et al. (2010)<br>Zhang et al. (2010)<br>Zhang et al. (2010)<br>Modarresi et al. (2007)<br>Kühne et al. (2005)<br>Kühne et al. (2005) | M<br>Q<br>Q<br>Q<br>Q<br>Q<br>Q<br>? | 12<br>287, 288<br>287, 289<br>287, 290<br>287, 291<br>67<br><br> |
| 2-methyl-4,6-dinitrophenol<br>$C_7H_6N_2O_5$<br>(6-methyl-2,4-dinitrophenol;<br>4,6-dinitro-$o$-cresol; DNOC)<br>[534-52-1]<br>ZXVONLUNISGICL-UHFFFAOYSA-N | $4.3\times10^1$<br>7.0<br>$9.2\times10^1$<br>$2.3\times10^1$<br>$9.1\times10^1$<br>$9.0\times10^{-1}$<br>7.0<br>1.7<br>$1.3\times10^3$<br>$1.5\times10^3$ | | Tremp et al. (1993)<br>Warner et al. (1980)<br>Mackay et al. (2006d)<br>Schwarzenbach et al. (1988)<br>Suntio et al. (1988)<br>Barcelo and Hennion (1997)<br>Shen (1982)<br>Keshavarz et al. (2022)<br>Duchowicz et al. (2020)<br>Wang et al. (2017) | M<br>M<br>V<br>V<br>V<br>X<br>C<br>Q<br>Q<br>Q | 12<br><br><br>12<br>12<br>567<br><br><br><br>80, 238 |



Table A4.8: Nitro compounds ($RNO_2$) (…continued)

| Substance Formula (Trivial Name) [CAS Registry Number] InChIKey | $H_s^{cp}$ (at $T^\ominus$) $\left[\dfrac{\text{mol}}{\text{m}^3\,\text{Pa}}\right]$ | $\dfrac{\text{d}\ln H_s^{cp}}{\text{d}(1/T)}$ [K] | Reference | Type | Note |
|---|---|---|---|---|---|
| | $1.7\times10^4$ | | Wang et al. (2017) | Q | 80, 239 |
| | 3.4 | | Wang et al. (2017) | Q | 80, 240 |
| | $3.2\times10^2$ | | Zhang et al. (2010) | Q | 287, 288 |
| | $2.3\times10^3$ | | Zhang et al. (2010) | Q | 287, 289 |
| | $1.9\times10^1$ | | Zhang et al. (2010) | Q | 287, 290 |
| | $7.2\times10^2$ | | Zhang et al. (2010) | Q | 287, 291 |
| | 4.1 | | Goodarzi et al. (2010) | Q | 568 |
| | $3.4\times10^1$ | | Modarresi et al. (2007) | Q | 67 |
| | | 5400 | Kühne et al. (2005) | Q | |
| | 7.0 | | Duchowicz et al. (2020) | ? | 185, 21 |
| | | 4200 | Kühne et al. (2005) | ? | |
| 5-nitrobenzimidazole $C_7H_5N_3O_2$ [94-52-0] XPAZGLFMMUODDK-UHFFFAOYSA-N | $2.7\times10^1$ | | HSDB (2015) | Q | 99 |
| 3-nitrobenzoic acid $C_7H_5NO_4$ [121-92-6] AFPHTEQTJZKQAQ-UHFFFAOYSA-N | $3.4\times10^3$ | | Abraham et al. (2019) | Q | |
| 4-nitrobenzoic acid $C_7H_5NO_4$ [62-23-7] OTLNPYWUJOZPPA-UHFFFAOYSA-N | $3.2\times10^3$ $2.6\times10^4$ | | Abraham et al. (2019) HSDB (2015) | Q Q | 99 |
| 3,5-dinitrobenzoic acid $C_7H_4N_2O_6$ [99-34-3] VYWYYJYRVSBHJQ-UHFFFAOYSA-N | $8.0\times10^4$ | | Abraham et al. (2019) | Q | |
| 2,4,6-trinitrobenzoic acid $C_7H_3N_3O_8$ [129-66-8] KAQBNBSMMVTKRN-UHFFFAOYSA-N | $3.8\times10^8$ | | HSDB (2015) | Q | 99 |
| dinitrotoluene $C_7H_6N_2O_4$ [25321-14-6] MOSFIJXAXDLOML-UHFFFAOYSA-N | $1.1\times10^2$ | | HSDB (2015) | Q | 447 |
| 1-methyl-3,5-dinitrobenzene $C_7H_6N_2O_4$ [618-85-9] RUIFULUFLANOCI-UHFFFAOYSA-N | $1.1\times10^2$ | | HSDB (2015) | Q | 447 |





Table A4.8: Nitro compounds ($RNO_2$) (...continued)

| Substance Formula (Trivial Name) [CAS Registry Number] InChIKey | $H_s^{cp}$ (at $T^{\ominus}$) $\left[\dfrac{\text{mol}}{\text{m}^3\,\text{Pa}}\right]$ | $\dfrac{\text{d}\ln H_s^{cp}}{\text{d}(1/T)}$ [K] | Reference | Type | Note |
|---|---|---|---|---|---|
| 1-methoxy-2-nitrobenzene $C_7H_7NO_3$ [91-23-6] CFBYEGUGFPZCNF-UHFFFAOYSA-N | $2.3\times10^1$ $2.3\times10^1$ $3.3\times10^1$ $1.2\times10^1$ $3.9\times10^{-1}$ $7.8$ $1.6\times10^2$ | | Duchowicz et al. (2020) HSDB (2015) Duchowicz et al. (2020) Raventos-Duran et al. (2010) Raventos-Duran et al. (2010) Raventos-Duran et al. (2010) Katritzky et al. (1998) | V V Q Q Q Q Q | 186 242, 243 244 245 |
| 2-methyl-5-nitrobenzenamine $C_7H_8N_2O_2$ (5-nitro-$o$-toluidine) [99-55-8] DSBIJCMXAIKKKI-UHFFFAOYSA-N | $1.2\times10^3$ | | HSDB (2015) | Q | 99 |
| 2-methoxy-5-nitrobenzenamine $C_7H_8N_2O_3$ (5-nitro-$o$-anisidine) [99-59-2] NIPDVSLAMPAWTP-UHFFFAOYSA-N | $7.6\times10^2$ | | HSDB (2015) | Q | 545 |
| (2-nitroethenyl)benzene $C_8H_7NO_2$ [102-96-5] PIAOLBVUVDXHHL-UHFFFAOYSA-N | $2.8$ | | HSDB (2015) | Q | 447 |
| 1,2-dimethyl-3-nitrobenzene $C_8H_9NO_2$ [83-41-0] FVHAWXWFPBPFOS-UHFFFAOYSA-N | $1.9\times10^{-1}$ $3.9\times10^{-1}$ $2.9\times10^{-1}$ $2.6\times10^{-1}$ $1.0\times10^{-1}$ | | HSDB (2015) Zhang et al. (2010) Zhang et al. (2010) Zhang et al. (2010) Zhang et al. (2010) | Q Q Q Q Q | 447 287, 288 287, 289 287, 290 287, 291 |
| 1,2-dimethyl-4-nitrobenzene $C_8H_9NO_2$ [99-51-4] HFZKOYWDLDYELC-UHFFFAOYSA-N | $3.9\times10^{-1}$ $3.1\times10^{-1}$ $8.0\times10^{-1}$ $1.0\times10^{-1}$ | | Zhang et al. (2010) Zhang et al. (2010) Zhang et al. (2010) Zhang et al. (2010) | Q Q Q Q | 287, 288 287, 289 287, 290 287, 291 |
| 1,4-dimethyl-2-nitrobenzene $C_8H_9NO_2$ [89-58-7] BSFHJMGROOFSRA-UHFFFAOYSA-N | $3.9\times10^{-1}$ $2.5\times10^{-1}$ $2.2\times10^{-1}$ $1.0\times10^{-1}$ | | Zhang et al. (2010) Zhang et al. (2010) Zhang et al. (2010) Zhang et al. (2010) | Q Q Q Q | 287, 288 287, 289 287, 290 287, 291 |
| 2,4-dimethyl-1-nitrobenzene $C_8H_9NO_2$ [89-87-2] BBUPBICWUURTNP-UHFFFAOYSA-N | $3.9\times10^{-1}$ $3.1\times10^{-1}$ $4.3\times10^{-1}$ $1.0\times10^{-1}$ | | Zhang et al. (2010) Zhang et al. (2010) Zhang et al. (2010) Zhang et al. (2010) | Q Q Q Q | 287, 288 287, 289 287, 290 287, 291 |
| 4-methyl-2-nitroanisole $C_8H_9NO_3$ [119-10-8] LGNMURXRPLMVJI-UHFFFAOYSA-N | $7.2$ $1.6$ $6.0\times10^1$ $2.7$ | | Zhang et al. (2010) Zhang et al. (2010) Zhang et al. (2010) Zhang et al. (2010) | Q Q Q Q | 287, 288 287, 289 287, 290 287, 291 |




Table A4.8: Nitro compounds ($RNO_2$) (...continued)

| Substance Formula (Trivial Name) [CAS Registry Number] InChIKey | $H_s^{cp}$ (at $T^\ominus$) $\left[\dfrac{\mathrm{mol}}{\mathrm{m^3\,Pa}}\right]$ | $\dfrac{\mathrm{d}\ln H_s^{cp}}{\mathrm{d}(1/T)}$ [K] | Reference | Type | Note |
|---|---|---|---|---|---|
| 2-methyl-3-nitrobenzoic acid $C_8H_7NO_4$ [1975-50-4] YPQAFWHSMWWPLX-UHFFFAOYSA-N | $2.2\times10^3$ | | Abraham et al. (2019) | Q | |
| 3-methyl-4-nitrobenzoic acid $C_8H_7NO_4$ [3113-71-1] XDTTUTIFWDAMIX-UHFFFAOYSA-N | $9.2\times10^2$ | | Abraham et al. (2019) | Q | |
| 2-methyl-3,5-dinitrobenzoic acid $C_8H_6N_2O_6$ [28169-46-2] CDVNZMKTJIBBBV-UHFFFAOYSA-N | $3.7\times10^6$ | | Abraham et al. (2019) | Q | |
| 1-(1-methylethyl)-4-nitrobenzene $C_9H_{11}NO_2$ [1817-47-6] JXMYUMNAEKRMIP-UHFFFAOYSA-N | $2.4\times10^{-1}$ $1.3\times10^{-1}$ $3.9\times10^{-1}$ $1.4\times10^{-1}$ | | Zhang et al. (2010) Zhang et al. (2010) Zhang et al. (2010) Zhang et al. (2010) | Q Q Q Q | 287, 288 287, 289 287, 290 287, 291 |
| 5-nitro-8-hydroxyquinoline $C_9H_6N_2O_3$ (nitroxoline) [4008-48-4] RJIWZDNTCBHXAL-UHFFFAOYSA-N | $2.7\times10^3$ | | Abraham et al. (2019) | Q | |
| 2-(1-methylpropyl)-4,6-dinitrophenol $C_{10}H_{12}N_2O_5$ (dinoseb) [88-85-7] OWZPCEFYPSAJFR-UHFFFAOYSA-N | 2.2 $2.0\times10^{-2}$ $1.9\times10^{-4}$ $1.4\times10^2$ $5.2\times10^2$ $1.3\times10^2$ $4.3\times10^2$ $6.2\times10^{-3}$ $1.7\times10^3$ | 6400 7200 | Tremp et al. (1993) Suntio et al. (1988) Barcelo and Hennion (1997) Zhang et al. (2010) Zhang et al. (2010) Zhang et al. (2010) Zhang et al. (2010) Goodarzi et al. (2010) Kühne et al. (2005) MacBean (2012a) Kühne et al. (2005) Mackay et al. (2006d) | M V X Q Q Q Q Q Q ? ? W | 12 12 567 287, 288 287, 289 287, 290 287, 291 568, 571 12 595 |
| 1-nitronaphthalene $C_{10}H_7NO_2$ [86-57-7] RJKGJBPXVHTNJL-UHFFFAOYSA-N | 4.6 5.6 $2.9\times10^{-1}$ 3.8 9.8 6.4 3.5 4.7 4.2 | | Chao et al. (2017) Altschuh et al. (1999) Mackay et al. (2006d) Mackay et al. (1995) Keshavarz et al. (2022) Duchowicz et al. (2020) Abraham et al. (2019) Parnis et al. (2015) Zhang et al. (2010) Zhang et al. (2010) | M M V V Q Q Q Q Q Q | 558 369 287, 288 287, 289 |



Table A4.8: Nitro compounds ($RNO_2$) (...continued)

| Substance Formula (Trivial Name) [CAS Registry Number] InChIKey | $H_s^{cp}$ (at $T^\ominus$) $\left[\dfrac{mol}{m^3\,Pa}\right]$ | $\dfrac{d\ln H_s^{cp}}{d(1/T)}$ [K] | Reference | Type | Note |
|---|---|---|---|---|---|
| | 1.6 | | Zhang et al. (2010) | Q | 287, 290 |
| | 4.7 | | Zhang et al. (2010) | Q | 287, 291 |
| | 2.1 | | Modarresi et al. (2007) | Q | 67 |
| | 5.6 | | Duchowicz et al. (2020) | ? | 185, 21 |
| 2-nitronaphthalene $C_{10}H_7NO_2$ [581-89-5] ZJYJZEAJZXVAMF-UHFFFAOYSA-N | 6.8 | | Parnis et al. (2015) | Q | 369 |
| 1,3-dinitronaphthalene $C_{10}H_6N_2O_4$ [606-37-1] ULALSFRIGPMWRS-UHFFFAOYSA-N | $1.2\times10^2$ | | Parnis et al. (2015) | Q | 369 |
| 1,5-dinitronaphthalene $C_{10}H_6N_2O_4$ [605-71-0] ZUTCJXFCHHDFJS-UHFFFAOYSA-N | $1.1\times10^2$ | | Parnis et al. (2015) | Q | 369 |
| 1,8-dinitronaphthalene $C_{10}H_6N_2O_4$ [602-38-0] AVCSMMMOCOTIHF-UHFFFAOYSA-N | $5.2\times10^3$ | | Parnis et al. (2015) | Q | 369 |
| dinoterb $C_{10}H_{12}N_2O_5$ [1420-07-1] IIPZYDQGBIWLBU-UHFFFAOYSA-N | 1.7 $9.1\times10^{-1}$ $9.3\times10^{-1}$ | | Barcelo and Hennion (1997) Goodarzi et al. (2010) MacBean (2012a) | X Q ? | 567 568, 569 |
| 4-(1-methylpropyl)-2-nitrophenol $C_{10}H_{13}NO_3$ (4-*sec*-butyl-2-nitrophenol) [3555-18-8] GCDCKEORRIGZKI-UHFFFAOYSA-N | $1.0\times10^{-1}$ $2.4\times10^{-1}$ $1.2\times10^{-1}$ $3.9\times10^{-1}$ $4.9\times10^{-1}$ | 5800 4300 | Tremp et al. (1993) Schwarzenbach et al. (1988) Raventos-Duran et al. (2010) Raventos-Duran et al. (2010) Raventos-Duran et al. (2010) Kühne et al. (2005) Kühne et al. (2005) | M V Q Q Q Q ? | 12 12 242, 243 244 245 |
| 1-methyl-4-nitronaphthalene $C_{11}H_9NO_2$ [880-93-3] FRLVKAJKOYFHKQ-UHFFFAOYSA-N | 7.7 | | Parnis et al. (2015) | Q | 369 |
| 1-methyl-5-nitronaphthalene $C_{11}H_9NO_2$ [91137-27-8] AOSRALZFDNFXFZ-UHFFFAOYSA-N | 5.0 | | Parnis et al. (2015) | Q | 369 |



Table A4.8: Nitro compounds ($RNO_2$) (...continued)

| Substance<br>Formula<br>(Trivial Name)<br>[CAS Registry Number]<br>InChIKey | $H_s^{cp}$<br>(at $T^{\ominus}$)<br>$\left[\dfrac{\text{mol}}{\text{m}^3\,\text{Pa}}\right]$ | $\dfrac{\text{d}\ln H_s^{cp}}{\text{d}(1/T)}$<br><br>[K] | Reference | Type | Note |
|---|---|---|---|---|---|
| 1-methyl-6-nitronaphthalene<br>$C_{11}H_9NO_2$<br>[105752-67-8]<br>SOXBGESONWKYDX-UHFFFAOYSA-N | 5.0 | | Parnis et al. (2015) | Q | 369 |
| 2-methyl-1-nitronaphthalene<br>$C_{11}H_9NO_2$<br>[881-03-8]<br>IZNWACYOILBFEG-UHFFFAOYSA-N | 1.5 | | Parnis et al. (2015) | Q | 369 |
| 3-methyl-1-nitronaphthalene<br>$C_{11}H_9NO_2$<br>[13615-38-8]<br>HKMFJWNUOWBRGF-UHFFFAOYSA-N | 3.9 | | Parnis et al. (2015) | Q | 369 |
| 3-nitrodibenzofuran<br>$C_{12}H_7NO_3$<br>[5410-97-9]<br>UVFAHDAUVZRVCC-UHFFFAOYSA-N | $2.6\times10^1$ | | Parnis et al. (2015) | Q | 369 |
| musk ambrette (artificial)<br>$C_{12}H_{16}N_2O_5$<br>[83-66-9]<br>SUAUILGSCPYJCS-UHFFFAOYSA-N | $1.4\times10^1$<br>$7.0\times10^2$<br>2.4<br>$2.2\times10^{-1}$<br>$4.6\times10^1$ | | HSDB (2015)<br>Zhang et al. (2010)<br>Zhang et al. (2010)<br>Zhang et al. (2010)<br>Zhang et al. (2010) | Q<br>Q<br>Q<br>Q<br>Q | 99<br>287, 288<br>287, 289<br>287, 290<br>287, 291 |
| bis($p$-nitrophenyl) ether<br>$C_{12}H_8N_2O_5$<br>[101-63-3]<br>MWAGUKZCDDRDCS-UHFFFAOYSA-N | $5.4\times10^3$<br>$2.3\times10^2$<br>$3.0\times10^3$<br>$1.1\times10^4$ | | Zhang et al. (2010)<br>Zhang et al. (2010)<br>Zhang et al. (2010)<br>Zhang et al. (2010) | Q<br>Q<br>Q<br>Q | 287, 288<br>287, 289<br>287, 290<br>287, 291 |
| 4-nitroazobenzene<br>$C_{12}H_9N_3O_2$<br>[2491-52-3]<br>TZTDJBMGPQLSLI-UHFFFAOYSA-N | $1.8\times10^1$ | | Ebert et al. (2023) | ? | 316 |
| 4-nitro-N-phenylbenzenamine<br>$C_{12}H_{10}N_2O_2$<br>[836-30-6]<br>XXYMSQQCBUKFHE-UHFFFAOYSA-N | $2.4\times10^3$<br>$1.7\times10^2$<br>$2.9\times10^4$<br>$2.5\times10^3$ | | Zhang et al. (2010)<br>Zhang et al. (2010)<br>Zhang et al. (2010)<br>Zhang et al. (2010) | Q<br>Q<br>Q<br>Q | 287, 288<br>287, 289<br>287, 290<br>287, 291 |
| 2-cyclohexyl-4,6-dinitrophenol<br>$C_{12}H_{14}N_2O_5$<br>[131-89-5]<br>QJYHUJAGJUHXJN-UHFFFAOYSA-N | $1.8\times10^2$ | | HSDB (2015) | Q | 99 |
| dinoseb acetate<br>$C_{12}H_{14}N_2O_6$<br>[2813-95-8]<br>RDJTWDKSYLLHRW-UHFFFAOYSA-N | $1.5\times10^3$ | | Ebert et al. (2023) | ? | 316 |



Table A4.8: Nitro compounds ($RNO_2$) (. . . continued)

| Substance Formula (Trivial Name) [CAS Registry Number] InChIKey | $H_s^{cp}$ (at $T^{\ominus}$) $\left[\dfrac{\text{mol}}{\text{m}^3\,\text{Pa}}\right]$ | $\dfrac{\text{d}\ln H_s^{cp}}{\text{d}(1/T)}$ [K] | Reference | Type | Note |
|---|---|---|---|---|---|
| dipicrylamine $C_{12}H_5N_7O_{12}$ (2,2',4,4',6,6'- hexanitrodiphenylamine) [131-73-7] CBCIHIVRDWLAME-UHFFFAOYSA-N | $4.3\times10^{11}$ | | HSDB (2015) | Q | 99 |
| 1,2-dihydro-5-nitroacenaphthylene $C_{12}H_9NO_2$ (5-nitroacenaphthene) [602-87-9] CUARLQDWYSRQDF-UHFFFAOYSA-N | 9.0 9.2 | | HSDB (2015) Parnis et al. (2015) | Q Q | 99 369 |
| 2-nitro-1,1'-biphenyl $C_{12}H_9NO_2$ [86-00-0] YOJKKXRJMXIKSR-UHFFFAOYSA-N | $1.6\times10^1$ | | Parnis et al. (2015) | Q | 369 |
| 3-nitro-1,1'-biphenyl $C_{12}H_9NO_2$ [2113-58-8] FYRPEHRWMVMHQM-UHFFFAOYSA-N | 9.9 | | Parnis et al. (2015) | Q | 369 |
| 4-nitro-1,1'-biphenyl $C_{12}H_9NO_2$ [92-93-3] BAJQRLZAPXASRD-UHFFFAOYSA-N | 2.8 $2.5\times10^1$ | | HSDB (2015) Parnis et al. (2015) | Q Q | 545 369 |
| 2-nitro-9H-fluorene $C_{13}H_9NO_2$ [607-57-8] XFOHWECQTFIEIX-UHFFFAOYSA-N | $3.4\times10^1$ $9.5\times10^1$ | | HSDB (2015) Parnis et al. (2015) | Q Q | 545 369 |
| 5-*tert*-butyl-4,6-dinitro-1,2,3-trimethylbenzene $C_{13}H_{18}N_2O_4$ [145-39-1] MINYPECWDZURGR-UHFFFAOYSA-N | $3.4\times10^1$ 2.1 $4.6\times10^{-2}$ 1.1 | | Zhang et al. (2010) Zhang et al. (2010) Zhang et al. (2010) Zhang et al. (2010) | Q Q Q Q | 287, 288 287, 289 287, 290 287, 291 |
| penoxaline $C_{13}H_{19}N_3O_4$ (pendimethalin) [40487-42-1] CHIFOSRWCNZCFN-UHFFFAOYSA-N | $1.2\times10^1$ $2.7\times10^{-1}$ $2.6\times10^{-3}$ $4.8\times10^{-3}$ 4.8 $3.4\times10^1$ $7.9\times10^{-1}$ | | Fendinger and Glotfelty (1990) Glotfelty et al. (1987) Barcelo and Hennion (1997) Goodarzi et al. (2010) Hilal et al. (2008) Modarresi et al. (2007) Maniere et al. (2011) | M V X Q Q Q ? | 567 568 67 165 |
| 2,6-dinitro-4-octylphenol $C_{14}H_{20}N_2O_5$ [4097-33-0] NYGISSDEOKKXOE-UHFFFAOYSA-N | $1.6\times10^4$ | | HSDB (2015) | Q | 99 |



Table A4.8: Nitro compounds ($RNO_2$) (... continued)

| Substance Formula (Trivial Name) [CAS Registry Number] InChIKey | $H_s^{cp}$ (at $T^\ominus$) $\left[\dfrac{\text{mol}}{\text{m}^3\,\text{Pa}}\right]$ | $\dfrac{\text{d}\ln H_s^{cp}}{\text{d}(1/T)}$ [K] | Reference | Type | Note |
|---|---|---|---|---|---|
| musk ketone $C_{14}H_{18}N_2O_5$ [81-14-1] WXCMHFPAUCOJIG-UHFFFAOYSA-N | 3.0 $5.2\times10^3$ $2.1\times10^4$ $2.6\times10^2$ 8.4 $5.0\times10^2$ | | Lee et al. (2012) HSDB (2015) Zhang et al. (2010) Zhang et al. (2010) Zhang et al. (2010) Zhang et al. (2010) | M Q Q Q Q Q | 99 287, 288 287, 289 287, 290 287, 291 |
| moskene $C_{14}H_{18}N_2O_4$ [116-66-5] UHWURQRPEIFIAK-UHFFFAOYSA-N | $4.8\times10^1$ $1.4\times10^1$ $7.5\times10^{-1}$ 2.5 | | Zhang et al. (2010) Zhang et al. (2010) Zhang et al. (2010) Zhang et al. (2010) | Q Q Q Q | 287, 288 287, 289 287, 290 287, 291 |
| nitrothal-isopropyl $C_{14}H_{17}NO_6$ [10552-74-6] VJAWBEFMCIINFU-UHFFFAOYSA-N | $5.7\times10^2$ | | Ebert et al. (2023) | ? | 316 |
| 9-ethyl-3-nitrocarbazole $C_{14}H_{12}N_2O_2$ [86-20-4] WONHLSYSHMRRGO-UHFFFAOYSA-N | $3.3\times10^2$ $6.9\times10^2$ $1.1\times10^3$ $2.5\times10^2$ | | Zhang et al. (2010) Zhang et al. (2010) Zhang et al. (2010) Zhang et al. (2010) | Q Q Q Q | 287, 288 287, 289 287, 290 287, 291 |
| 2-nitroanthracene $C_{14}H_9NO_2$ [3586-69-4] NZWGQBVOKHEKLD-UHFFFAOYSA-N | $1.0\times10^2$ | | Parnis et al. (2015) | Q | 369 |
| 9-nitroanthracene $C_{14}H_9NO_2$ [602-60-8] LSIKFJXEYJIZNB-UHFFFAOYSA-N | $1.6\times10^1$ | | Parnis et al. (2015) | Q | 369 |
| 3-nitrophenanthrene $C_{14}H_9NO_2$ [17024-19-0] CPRHWWUDRYJODK-UHFFFAOYSA-N | $5.4\times10^1$ | | Parnis et al. (2015) | Q | 369 |
| 9-nitrophenanthrene $C_{14}H_9NO_2$ [954-46-1] QTTCNQHPKFAYEZ-UHFFFAOYSA-N | $3.2\times10^1$ | | Parnis et al. (2015) | Q | 369 |
| binapacryl $C_{15}H_{18}N_2O_6$ [485-31-4] ZRDUSMYWDRPZRM-UHFFFAOYSA-N | $2.2\times10^2$ | | Ebert et al. (2023) | ? | 316 |
| 2-nitrofluoranthene $C_{16}H_9NO_2$ [13177-29-2] VBCBFNMZBHKVQN-UHFFFAOYSA-N | $8.2\times10^1$ | | Parnis et al. (2015) | Q | 369 |





Table A4.8: Nitro compounds ($RNO_2$) (...continued)

| Substance Formula (Trivial Name) [CAS Registry Number] InChIKey | $H_s^{cp}$ (at $T^\ominus$) $\left[\dfrac{mol}{m^3\,Pa}\right]$ | $\dfrac{d\ln H_s^{cp}}{d(1/T)}$ [K] | Reference | Type | Note |
|---|---|---|---|---|---|
| 3-nitrofluoranthene $C_{16}H_9NO_2$ [892-21-7] PIHGQKMEAMSUNA-UHFFFAOYSA-N | $1.1\times10^2$ | | Parnis et al. (2015) | Q | 369 |
| 1-[(2,4-dinitrophenyl)azo]-2-naphthol $C_{16}H_{10}N_4O_5$ (C.I. pigment orange 5) [3468-63-1] HBHZKFOUIUMKHV-UHFFFAOYSA-N | $1.1\times10^9$ | | HSDB (2015) | Q | 99 |
| 3,7-dinitrofluoranthene $C_{16}H_8N_2O_4$ [105735-71-5] WAAHHGKGQYVTNS-UHFFFAOYSA-N | $4.9\times10^4$ | | HSDB (2015) | Q | 99 |
| 1,3-dinitropyrene $C_{16}H_8N_2O_4$ [75321-20-9] UJIPQBOHUQDIAA-UHFFFAOYSA-N | $6.3\times10^3$ | | Parnis et al. (2015) | Q | 369 |
| 1,6-dinitropyrene $C_{16}H_8N_2O_4$ [42397-64-8] GUXACCKTQWVTLG-UHFFFAOYSA-N | $7.6\times10^4$ $7.3\times10^3$ | | HSDB (2015) Parnis et al. (2015) | Q Q | 99 369 |
| 1,8-dinitropyrene $C_{16}H_8N_2O_4$ [42397-65-9] BLYXNIHKOMELAP-UHFFFAOYSA-N | $7.6\times10^4$ $2.3\times10^4$ | | HSDB (2015) Parnis et al. (2015) | Q Q | 99 369 |
| 1-nitropyrene $C_{16}H_9NO_2$ [5522-43-0] ALRLPDGCPYIVHP-UHFFFAOYSA-N | $3.9\times10^2$ $2.0\times10^2$ | | HSDB (2015) Parnis et al. (2015) | Q Q | 99 369 |
| 2-nitropyrene $C_{16}H_9NO_2$ [789-07-1] MAZCGYFIOOIVHE-UHFFFAOYSA-N | $9.0\times10^1$ | | Parnis et al. (2015) | Q | 369 |
| 4-nitropyrene $C_{16}H_9NO_2$ [57835-92-4] UISKIUIWPSPSAV-UHFFFAOYSA-N | $3.9\times10^2$ | | HSDB (2015) | Q | 99 |





Table A4.8: Nitro compounds ($RNO_2$) (... continued)

| Substance<br>Formula<br>(Trivial Name)<br>[CAS Registry Number]<br>InChIKey | $H_s^{cp}$<br>(at $T^\ominus$)<br>$\left[\dfrac{mol}{m^3\,Pa}\right]$ | $\dfrac{d\ln H_s^{cp}}{d(1/T)}$<br><br>[K] | Reference | Type | Note |
|---|---|---|---|---|---|
| N,N-diethyl-4-[(4-nitrophenyl)azo]aniline<br>$C_{16}H_{18}N_4O_2$<br>[3025-52-3]<br>LVQIWDUSUJTZJF-ISLYRVAYSA-N | 3.7<br><br>$1.5\times10^3$ | | Duchowicz et al. (2020)<br><br>Duchowicz et al. (2020) | V<br><br>Q | 186 |
| disperse red 1<br>$C_{16}H_{18}N_4O_3$<br>[2872-52-8]<br>FOQABOMYTOFLPZ-ZCXUNETKSA-N | $1.2\times10^8$<br>$1.1\times10^6$ | | Duchowicz et al. (2020)<br>Duchowicz et al. (2020) | V<br>Q | 186 |
| 1-[(4-methyl-2-nitrophenyl)azo]-2-naphthalenol<br>$C_{17}H_{13}N_3O_3$<br>(C.I. Pigment Red 3)<br>[2425-85-6]<br>ZLFVRXUOSPRRKQ-UHFFFAOYSA-N | $8.2\times10^6$ | | HSDB (2015) | Q | 99 |
| phenyl 1-hydroxy-4-nitro-2-naphthoate<br>$C_{17}H_{11}NO_5$<br>[65208-34-6]<br>DMPUGHYNLCGVPX-UHFFFAOYSA-N | $1.5\times10^4$<br><br>$6.7\times10^5$<br>$1.1\times10^2$<br>$2.7\times10^5$ | | Zhang et al. (2010)<br><br>Zhang et al. (2010)<br>Zhang et al. (2010)<br>Zhang et al. (2010) | Q<br><br>Q<br>Q<br>Q | 287, 288<br><br>287, 289<br>287, 290<br>287, 291 |
| 7-nitrobenz[$a$]anthracene<br>$C_{18}H_{11}NO_2$<br>[20268-51-3]<br>KOPVBVBUIYTJBG-UHFFFAOYSA-N | $9.9\times10^1$ | | Parnis et al. (2015) | Q | 369 |
| 6-nitrochrysene<br>$C_{18}H_{11}NO_2$<br>[7496-02-8]<br>UAWLTQJFZUYROA-UHFFFAOYSA-N | $6.6\times10^2$ | | HSDB (2015) | Q | 99 |
| meptyldinocap<br>$C_{18}H_{24}N_2O_6$<br>[131-72-6]<br>NIOPZPCMRQGZCE-WEVVVXLNSA-N | $8.6\times10^1$ | | Maniere et al. (2011) | ? | 241, 165 |
| 1-nitrobenzo[$a$]pyrene<br>$C_{20}H_{11}NO_2$<br>[70021-99-7]<br>ICKISBPYFVBVQG-UHFFFAOYSA-N | $3.1\times10^3$ | | HSDB (2015) | Q | 447 |
| 3-nitrobenzo[$a$]pyrene<br>$C_{20}H_{11}NO_2$<br>[70021-98-6]<br>CQIJHYPCKYZMIV-UHFFFAOYSA-N | $3.1\times10^3$ | | HSDB (2015) | Q | 447 |





Table A4.8: Nitro compounds ($RNO_2$) (...continued)

| Substance / Formula / (Trivial Name) / [CAS Registry Number] / InChIKey | $H_s^{cp}$ (at $T^\ominus$) $\left[\dfrac{\text{mol}}{\text{m}^3\,\text{Pa}}\right]$ | $\dfrac{\mathrm{d}\ln H_s^{cp}}{\mathrm{d}(1/T)}$ [K] | Reference | Type | Note |
|---|---|---|---|---|---|
| 6-nitrobenzo[$a$]pyrene $C_{20}H_{11}NO_2$ [63041-90-7] NMMAFYSZGOFZCM-UHFFFAOYSA-N | $3.1\times10^3$ $3.3\times10^2$ | | HSDB (2015) Parnis et al. (2015) | Q Q | 447 369 |
| MCM:NPHEN1OOH $C_6H_5NO_4$ QKMAXSKQIHICGE-UHFFFAOYSA-N | $2.8\times10^4$ $3.0\times10^4$ $3.3\times10^3$ | | Wang et al. (2017) Wang et al. (2017) Wang et al. (2017) | Q Q Q | 80, 238 80, 239 80, 240 |
| MCM:NCATECHOL $C_6H_5NO_4$ XJNPNXSISMKQEX-UHFFFAOYSA-N | $3.2\times10^7$ $8.0\times10^7$ $6.0\times10^6$ | | Wang et al. (2017) Wang et al. (2017) Wang et al. (2017) | Q Q Q | 80, 238 80, 239 80, 240 |
| MCM:NCRES1OOH $C_7H_7NO_4$ PUXUHFKGXFXGAC-UHFFFAOYSA-N | $1.7\times10^4$ $8.9\times10^3$ $4.7$ | | Wang et al. (2017) Wang et al. (2017) Wang et al. (2017) | Q Q Q | 80, 238 80, 239 80, 240 |
| MCM:MNCATECH $C_7H_7NO_4$ DPKDSDOOIONLAG-UHFFFAOYSA-N | $5.6\times10^3$ $8.5\times10^5$ $3.6$ | | Wang et al. (2017) Wang et al. (2017) Wang et al. (2017) | Q Q Q | 80, 238 80, 239 80, 240 |
| MCM:NEBNZ1OOH $C_8H_9NO_4$ WELXCAQABDDCET-UHFFFAOYSA-N | $1.3\times10^4$ $4.9\times10^3$ $1.7$ | | Wang et al. (2017) Wang et al. (2017) Wang et al. (2017) | Q Q Q | 80, 238 80, 239 80, 240 |
| MCM:NMXYOL1OOH $C_8H_9NO_4$ WHVCHTZZXIECJY-UHFFFAOYSA-N | $9.6\times10^3$ $2.2\times10^4$ $2.7$ | | Wang et al. (2017) Wang et al. (2017) Wang et al. (2017) | Q Q Q | 80, 238 80, 239 80, 240 |
| MCM:NOXYOL1OOH $C_8H_9NO_4$ WTJTXPQAQHBAPV-UHFFFAOYSA-N | $9.6\times10^3$ $1.2\times10^4$ $8.9$ | | Wang et al. (2017) Wang et al. (2017) Wang et al. (2017) | Q Q Q | 80, 238 80, 239 80, 240 |
| MCM:NPXYOL1OOH $C_8H_9NO_4$ MKCNKDVBVLGUFV-UHFFFAOYSA-N | $9.6\times10^3$ $1.2\times10^4$ $2.7$ | | Wang et al. (2017) Wang et al. (2017) Wang et al. (2017) | Q Q Q | 80, 238 80, 239 80, 240 |
| MCM:DM124OHNO2 $C_8H_9NO_3$ KGDIYDUZVHFMHQ-UHFFFAOYSA-N | $2.3$ $2.4$ $5.6\times10^{-1}$ | | Wang et al. (2017) Wang et al. (2017) Wang et al. (2017) | Q Q Q | 80, 238 80, 239 80, 240 |
| MCM:DMPHOHNO2 $C_8H_9NO_3$ YXNYMZXPUWOUJT-UHFFFAOYSA-N | $2.3$ $1.2\times10^1$ $3.5\times10^{-1}$ | | Wang et al. (2017) Wang et al. (2017) Wang et al. (2017) | Q Q Q | 80, 238 80, 239 80, 240 |
| MCM:DNEBNZOL $C_8H_8N_2O_5$ SYWMIOFIFBKHTK-UHFFFAOYSA-N | $1.2\times10^3$ $9.6\times10^3$ $2.1$ | | Wang et al. (2017) Wang et al. (2017) Wang et al. (2017) | Q Q Q | 80, 238 80, 239 80, 240 |
| MCM:DNMXYOL $C_8H_8N_2O_5$ MHXAYMPVZDCVJX-UHFFFAOYSA-N | $1.1\times10^3$ $2.1\times10^4$ $1.9\times10^{-1}$ | | Wang et al. (2017) Wang et al. (2017) Wang et al. (2017) | Q Q Q | 80, 238 80, 239 80, 240 |




Table A4.8: Nitro compounds ($RNO_2$) (...continued)

| Substance<br>Formula<br>(Trivial Name)<br>[CAS Registry Number]<br>InChIKey | $H_s^{cp}$ (at $T^\ominus$)<br>$\left[\dfrac{\text{mol}}{\text{m}^3\,\text{Pa}}\right]$ | $\dfrac{\text{d}\ln H_s^{cp}}{\text{d}(1/T)}$<br>[K] | Reference | Type | Note |
|---|---|---|---|---|---|
| MCM:DNOXYOL<br>$C_8H_8N_2O_5$<br>JCTQXQRGEDMCSI-UHFFFAOYSA-N | $8.9\times10^2$<br>$1.6\times10^4$<br>2.6 | | Wang et al. (2017)<br>Wang et al. (2017)<br>Wang et al. (2017) | Q<br>Q<br>Q | 80, 238<br>80, 239<br>80, 240 |
| MCM:DNPXYOL<br>$C_8H_8N_2O_5$<br>RROXWBBJMPCPHD-UHFFFAOYSA-N | $8.9\times10^2$<br>$6.5\times10^3$<br>1.4 | | Wang et al. (2017)<br>Wang et al. (2017)<br>Wang et al. (2017) | Q<br>Q<br>Q | 80, 238<br>80, 239<br>80, 240 |
| MCM:EBNZOHNO2<br>$C_8H_9NO_3$<br>RSETVJMGQZKFCF-UHFFFAOYSA-N | 3.5<br>$1.3\times10^1$<br>$7.3\times10^{-2}$ | | Wang et al. (2017)<br>Wang et al. (2017)<br>Wang et al. (2017) | Q<br>Q<br>Q | 80, 238<br>80, 239<br>80, 240 |
| MCM:ENCATECH<br>$C_8H_9NO_4$<br>CGDBCFCRALNPBR-UHFFFAOYSA-N | $4.5\times10^3$<br>$5.1\times10^5$<br>2.0 | | Wang et al. (2017)<br>Wang et al. (2017)<br>Wang et al. (2017) | Q<br>Q<br>Q | 80, 238<br>80, 239<br>80, 240 |
| MCM:MXNCATECH<br>$C_8H_9NO_4$<br>DNJZITGVNONOFW-UHFFFAOYSA-N | $3.3\times10^3$<br>$4.1\times10^5$<br>2.0 | | Wang et al. (2017)<br>Wang et al. (2017)<br>Wang et al. (2017) | Q<br>Q<br>Q | 80, 238<br>80, 239<br>80, 240 |
| MCM:MXY1OHNO2<br>$C_8H_9NO_3$<br>KJRCHILWKQLEBC-UHFFFAOYSA-N | 2.3<br>$1.7\times10^1$<br>$1.4\times10^{-1}$ | | Wang et al. (2017)<br>Wang et al. (2017)<br>Wang et al. (2017) | Q<br>Q<br>Q | 80, 238<br>80, 239<br>80, 240 |
| MCM:OXNCATECH<br>$C_8H_9NO_4$<br>BBBHATWMUPCFKG-UHFFFAOYSA-N | $3.3\times10^3$<br>$8.0\times10^5$<br>3.1 | | Wang et al. (2017)<br>Wang et al. (2017)<br>Wang et al. (2017) | Q<br>Q<br>Q | 80, 238<br>80, 239<br>80, 240 |
| MCM:OXY1OHNO2<br>$C_8H_9NO_3$<br>KXWOAPZXQJGYPU-UHFFFAOYSA-N | 2.3<br>$2.6\times10^1$<br>$2.0\times10^{-1}$ | | Wang et al. (2017)<br>Wang et al. (2017)<br>Wang et al. (2017) | Q<br>Q<br>Q | 80, 238<br>80, 239<br>80, 240 |
| MCM:PXNCATECH<br>$C_8H_9NO_4$<br>MAHUOFZDOBGDSU-UHFFFAOYSA-N | $1.1\times10^7$<br>$2.5\times10^8$<br>$4.7\times10^4$ | | Wang et al. (2017)<br>Wang et al. (2017)<br>Wang et al. (2017) | Q<br>Q<br>Q | 80, 238<br>80, 239<br>80, 240 |
| MCM:PXY1OHNO2<br>$C_8H_9NO_3$<br>VIQHHRZADKSPIM-UHFFFAOYSA-N | 2.3<br>8.7<br>$1.0\times10^{-1}$ | | Wang et al. (2017)<br>Wang et al. (2017)<br>Wang et al. (2017) | Q<br>Q<br>Q | 80, 238<br>80, 239<br>80, 240 |
| MCM:NIPBNZ1OOH<br>$C_9H_{11}NO_4$<br>IXEXRYHUTBEDFT-UHFFFAOYSA-N | $1.2\times10^4$<br>$3.6\times10^3$<br>1.2 | | Wang et al. (2017)<br>Wang et al. (2017)<br>Wang et al. (2017) | Q<br>Q<br>Q | 80, 238<br>80, 239<br>80, 240 |
| MCM:NMETOL1OOH<br>$C_9H_{11}NO_4$<br>OTBVEKBEKVMFBF-UHFFFAOYSA-N | $8.0\times10^3$<br>$4.4\times10^3$<br>2.1 | | Wang et al. (2017)<br>Wang et al. (2017)<br>Wang et al. (2017) | Q<br>Q<br>Q | 80, 238<br>80, 239<br>80, 240 |
| MCM:NOETOL1OOH<br>$C_9H_{11}NO_4$<br>ZYYASKSQLLYACE-UHFFFAOYSA-N | $8.0\times10^3$<br>$6.5\times10^3$<br>3.2 | | Wang et al. (2017)<br>Wang et al. (2017)<br>Wang et al. (2017) | Q<br>Q<br>Q | 80, 238<br>80, 239<br>80, 240 |



Table A4.8: Nitro compounds ($RNO_2$) (... continued)

| Substance<br>Formula<br>(Trivial Name)<br>[CAS Registry Number]<br>InChIKey | $H_s^{cp}$<br>(at $T^{\ominus}$)<br>$\left[\dfrac{\text{mol}}{\text{m}^3\,\text{Pa}}\right]$ | $\dfrac{\text{d}\ln H_s^{cp}}{\text{d}(1/T)}$<br><br>[K] | Reference | Type | Note |
|---|---|---|---|---|---|
| MCM:NPBNZ1OOH<br>$C_9H_{11}NO_4$<br>CVKCTJQZVRPANA-UHFFFAOYSA-N | $1.2\times10^4$<br>$3.6\times10^3$<br>$1.5$ | | Wang et al. (2017)<br>Wang et al. (2017)<br>Wang et al. (2017) | Q<br>Q<br>Q | 80, 238<br>80, 239<br>80, 240 |
| MCM:NPETOL1OOH<br>$C_9H_{11}NO_4$<br>PGRGKKYGDUKVIK-UHFFFAOYSA-N | $8.0\times10^3$<br>$3.2\times10^3$<br>$2.0$ | | Wang et al. (2017)<br>Wang et al. (2017)<br>Wang et al. (2017) | Q<br>Q<br>Q | 80, 238<br>80, 239<br>80, 240 |
| MCM:NT123L1OOH<br>$C_9H_{11}NO_4$<br>QUQYRHULCDEZAA-UHFFFAOYSA-N | $5.6\times10^3$<br>$1.4\times10^4$<br>$1.4\times10^1$ | | Wang et al. (2017)<br>Wang et al. (2017)<br>Wang et al. (2017) | Q<br>Q<br>Q | 80, 238<br>80, 239<br>80, 240 |
| MCM:NT124L1OOH<br>$C_9H_{11}NO_4$<br>DLQITMPSPXAOFG-UHFFFAOYSA-N | $5.6\times10^3$<br>$4.6\times10^4$<br>$2.9\times10^2$ | | Wang et al. (2017)<br>Wang et al. (2017)<br>Wang et al. (2017) | Q<br>Q<br>Q | 80, 238<br>80, 239<br>80, 240 |
| MCM:DNIPBNZOL<br>$C_9H_{10}N_2O_5$<br>[29385-11-3]<br>HBYHYLBZPLCIEE-UHFFFAOYSA-N | $1.1\times10^3$<br>$3.6\times10^3$<br>$1.4$<br>$2.5\times10^1$<br>$6.2\times10^2$<br>$1.6\times10^2$ | | Wang et al. (2017)<br>Wang et al. (2017)<br>Wang et al. (2017)<br>Raventos-Duran et al. (2010)<br>Raventos-Duran et al. (2010)<br>Raventos-Duran et al. (2010) | Q<br>Q<br>Q<br>Q<br>Q<br>Q | 80, 238<br>80, 239<br>80, 240<br>271, 243<br>244<br>245 |
| MCM:DNMETOL<br>$C_9H_{10}N_2O_5$<br>NLLBSWNZAUXBGJ-UHFFFAOYSA-N | $8.7\times10^2$<br>$2.4\times10^4$<br>$1.3\times10^{-1}$ | | Wang et al. (2017)<br>Wang et al. (2017)<br>Wang et al. (2017) | Q<br>Q<br>Q | 80, 238<br>80, 239<br>80, 240 |
| MCM:DNOETOL<br>$C_9H_{10}N_2O_5$<br>AGMONASRVOHMRW-UHFFFAOYSA-N | $7.3\times10^2$<br>$1.0\times10^4$<br>$1.9$ | | Wang et al. (2017)<br>Wang et al. (2017)<br>Wang et al. (2017) | Q<br>Q<br>Q | 80, 238<br>80, 239<br>80, 240 |
| MCM:DNPBNZOL<br>$C_9H_{10}N_2O_5$<br>FCIYPWNHZQHVEQ-UHFFFAOYSA-N | $1.1\times10^3$<br>$5.8\times10^3$<br>$1.7$ | | Wang et al. (2017)<br>Wang et al. (2017)<br>Wang et al. (2017) | Q<br>Q<br>Q | 80, 238<br>80, 239<br>80, 240 |
| MCM:DNPETOL<br>$C_9H_{10}N_2O_5$<br>DZWALJMRDDSGJQ-UHFFFAOYSA-N | $7.3\times10^2$<br>$3.9\times10^3$<br>$8.9\times10^{-1}$ | | Wang et al. (2017)<br>Wang et al. (2017)<br>Wang et al. (2017) | Q<br>Q<br>Q | 80, 238<br>80, 239<br>80, 240 |
| MCM:DNT123BOL<br>$C_9H_{10}N_2O_5$<br>KTYJIHPJHVSEQH-UHFFFAOYSA-N | $6.2\times10^2$<br>$1.7\times10^4$<br>$3.8$ | | Wang et al. (2017)<br>Wang et al. (2017)<br>Wang et al. (2017) | Q<br>Q<br>Q | 80, 238<br>80, 239<br>80, 240 |
| MCM:DNT124BOL<br>$C_9H_{10}N_2O_5$<br>HCZZKJMHIZGVJN-UHFFFAOYSA-N | $4.0\times10^5$<br>$1.4\times10^4$<br>$3.9\times10^3$ | | Wang et al. (2017)<br>Wang et al. (2017)<br>Wang et al. (2017) | Q<br>Q<br>Q | 80, 238<br>80, 239<br>80, 240 |
| MCM:HOEMPHNO2<br>$C_9H_{11}NO_3$<br>YSPMBFYZTNTHGD-UHFFFAOYSA-N | $2.1$<br>$1.0\times10^1$<br>$2.6\times10^{-1}$ | | Wang et al. (2017)<br>Wang et al. (2017)<br>Wang et al. (2017) | Q<br>Q<br>Q | 80, 238<br>80, 239<br>80, 240 |



Table A4.8: Nitro compounds ($RNO_2$) (...continued)

| Substance<br>Formula<br>(Trivial Name)<br>[CAS Registry Number]<br>InChIKey | $H_s^{cp}$<br>(at $T^{\ominus}$)<br>$\left[\dfrac{\mathrm{mol}}{\mathrm{m^3\,Pa}}\right]$ | $\dfrac{\mathrm{d}\ln H_s^{cp}}{\mathrm{d}(1/T)}$<br><br>[K] | Reference | Type | Note |
|---|---|---|---|---|---|
| MCM:IPBNZOHNO2 | 3.2 | | Wang et al. (2017) | Q | 80, 238 |
| $C_9H_{11}NO_3$ | 5.6 | | Wang et al. (2017) | Q | 80, 239 |
| RRFSVDKJKYCCEK-UHFFFAOYSA-N | $6.5 \times 10^{-2}$ | | Wang et al. (2017) | Q | 80, 240 |
| MCM:IPNCATECH | $4.2 \times 10^3$ | | Wang et al. (2017) | Q | 80, 238 |
| $C_9H_{11}NO_4$ | $2.0 \times 10^5$ | | Wang et al. (2017) | Q | 80, 239 |
| MCPZUPZVJHDAPC-UHFFFAOYSA-N | 1.6 | | Wang et al. (2017) | Q | 80, 240 |
| MCM:MET1OHNO2 | 2.1 | | Wang et al. (2017) | Q | 80, 238 |
| $C_9H_{11}NO_3$ | $1.0 \times 10^1$ | | Wang et al. (2017) | Q | 80, 239 |
| QDGFKFKXXCXYOS-UHFFFAOYSA-N | $8.7 \times 10^{-2}$ | | Wang et al. (2017) | Q | 80, 240 |
| MCM:MTNCATECH | $2.6 \times 10^3$ | | Wang et al. (2017) | Q | 80, 238 |
| $C_9H_{11}NO_4$ | $2.6 \times 10^5$ | | Wang et al. (2017) | Q | 80, 239 |
| CJFAOEWQLMJWBQ-UHFFFAOYSA-N | 1.0 | | Wang et al. (2017) | Q | 80, 240 |
| MCM:OET1OHNO2 | 2.1 | | Wang et al. (2017) | Q | 80, 238 |
| $C_9H_{11}NO_3$ | $1.5 \times 10^1$ | | Wang et al. (2017) | Q | 80, 239 |
| MEMHWZGPMLWUCR-UHFFFAOYSA-N | $1.4 \times 10^{-1}$ | | Wang et al. (2017) | Q | 80, 240 |
| MCM:OTNCATECH | $2.6 \times 10^3$ | | Wang et al. (2017) | Q | 80, 238 |
| $C_9H_{11}NO_4$ | $4.6 \times 10^5$ | | Wang et al. (2017) | Q | 80, 239 |
| JWEUFFGZFSJOFB-UHFFFAOYSA-N | 2.2 | | Wang et al. (2017) | Q | 80, 240 |
| MCM:PBNZOHNO2 | 2.8 | | Wang et al. (2017) | Q | 80, 238 |
| $C_9H_{11}NO_3$ | 8.5 | | Wang et al. (2017) | Q | 80, 239 |
| SNIBPLNVBYUZKZ-UHFFFAOYSA-N | $6.3 \times 10^{-2}$ | | Wang et al. (2017) | Q | 80, 240 |
| MCM:PET1OHNO2 | 2.1 | | Wang et al. (2017) | Q | 80, 238 |
| $C_9H_{11}NO_3$ | 5.6 | | Wang et al. (2017) | Q | 80, 239 |
| DXOFURYTWRHSOS-UHFFFAOYSA-N | $6.3 \times 10^{-2}$ | | Wang et al. (2017) | Q | 80, 240 |
| MCM:PNCATECH | $3.5 \times 10^3$ | | Wang et al. (2017) | Q | 80, 238 |
| $C_9H_{11}NO_4$ | $2.9 \times 10^5$ | | Wang et al. (2017) | Q | 80, 239 |
| UWDKGIAMIWLDOZ-UHFFFAOYSA-N | 1.7 | | Wang et al. (2017) | Q | 80, 240 |
| MCM:PTNCATECH | $8.9 \times 10^6$ | | Wang et al. (2017) | Q | 80, 238 |
| $C_9H_{11}NO_4$ | $1.8 \times 10^8$ | | Wang et al. (2017) | Q | 80, 239 |
| POSTUROPYARVPT-UHFFFAOYSA-N | $3.7 \times 10^4$ | | Wang et al. (2017) | Q | 80, 240 |
| MCM:T123NCATEC | $2.0 \times 10^3$ | | Wang et al. (2017) | Q | 80, 238 |
| $C_9H_{11}NO_4$ | $6.0 \times 10^5$ | | Wang et al. (2017) | Q | 80, 239 |
| NRIGBHNXEXAXSY-UHFFFAOYSA-N | 2.0 | | Wang et al. (2017) | Q | 80, 240 |
| MCM:T124NCATEC | $6.6 \times 10^6$ | | Wang et al. (2017) | Q | 80, 238 |
| $C_9H_{11}NO_4$ | $4.8 \times 10^8$ | | Wang et al. (2017) | Q | 80, 239 |
| XNILZTOSMSXQOP-UHFFFAOYSA-N | $8.7 \times 10^3$ | | Wang et al. (2017) | Q | 80, 240 |
| MCM:TM123OHNO2 | 1.5 | | Wang et al. (2017) | Q | 80, 238 |
| $C_9H_{11}NO_3$ | $3.3 \times 10^1$ | | Wang et al. (2017) | Q | 80, 239 |
| GRYTUHBAWMNFLK-UHFFFAOYSA-N | $3.0 \times 10^{-1}$ | | Wang et al. (2017) | Q | 80, 240 |





Table A4.8: Nitro compounds ($RNO_2$) (...continued)

| Substance Formula (Trivial Name) [CAS Registry Number] InChIKey | $H_s^{cp}$ (at $T^\ominus$) $\left[\dfrac{\mathrm{mol}}{\mathrm{m^3\,Pa}}\right]$ | $\dfrac{\mathrm{d}\ln H_s^{cp}}{\mathrm{d}(1/T)}$ [K] | Reference | Type | Note |
|---|---|---|---|---|---|
| MCM:TM124OHNO2 | $9.6\times10^2$ | | Wang et al. (2017) | Q | 80, 238 |
| $C_9H_{11}NO_3$ | $2.4\times10^4$ | | Wang et al. (2017) | Q | 80, 239 |
| NKOCMNXLDSIDQC-UHFFFAOYSA-N | $1.0\times10^3$ | | Wang et al. (2017) | Q | 80, 240 |





## A5   Organic species with fluorine (F)

### A5.1   Organic fluorine

Table A5.1: Organic fluorine

| Substance Formula (Trivial Name) [CAS Registry Number] InChIKey | $H_s^{cp}$ (at $T^{\ominus}$) $\left[\dfrac{\text{mol}}{\text{m}^3\,\text{Pa}}\right]$ | $\dfrac{\text{d}\ln H_s^{cp}}{\text{d}(1/T)}$ [K] | Reference | Type | Note |
|---|---|---|---|---|---|
| fluoromethane | $6.1\times10^{-4}$ | 2100 | Burkholder et al. (2019) | L | 1 |
| $CH_3F$ | $6.1\times10^{-4}$ | 2000 | Burkholder et al. (2015) | L | |
| [593-53-3] | $6.2\times10^{-4}$ | 2200 | Brockbank (2013) | L | 1, 596 |
| NBVXSUQYWXRMNV-UHFFFAOYSA-N | $6.1\times10^{-4}$ | 2000 | Sander et al. (2011) | L | |
| | $6.1\times10^{-4}$ | 2000 | Sander et al. (2006) | L | |
| | $5.8\times10^{-4}$ | 2200 | Wilhelm et al. (1977) | L | |
| | $5.8\times10^{-4}$ | 2100 | Swain and Thornton (1962) | M | |
| | $5.8\times10^{-4}$ | 2200 | Glew and Moelwyn-Hughes (1953) | M | 597 |
| | $5.8\times10^{-4}$ | | Duchowicz et al. (2020) | V | 186 |
| | $5.1\times10^{-4}$ | | Mackay and Shiu (1981) | V | |
| | $5.8\times10^{-4}$ | | Hine and Mookerjee (1975) | V | |
| | $7.1\times10^{-4}$ | | Yaws (2003) | X | 237, 80 |
| | $1.0\times10^{-7}$ | | Hayer et al. (2022) | Q | 20 |
| | $2.6\times10^{-3}$ | | Duchowicz et al. (2020) | Q | |
| | $8.7\times10^{-5}$ | | Gharagheizi et al. (2012) | Q | |
| | $4.9\times10^{-4}$ | | Raventos-Duran et al. (2010) | Q | 242, 243 |
| | $6.2\times10^{-4}$ | | Raventos-Duran et al. (2010) | Q | 244 |
| | $6.2\times10^{-4}$ | | Raventos-Duran et al. (2010) | Q | 245 |
| | $6.7\times10^{-4}$ | | Gharagheizi et al. (2010) | Q | 246 |
| | $9.2\times10^{-5}$ | | Hilal et al. (2008) | Q | |
| | | 2200 | Kühne et al. (2005) | Q | |
| | $5.1\times10^{-4}$ | | Yaffe et al. (2003) | Q | 248, 249 |
| | $7.7\times10^{-4}$ | | English and Carroll (2001) | Q | 230, 231 |
| | $6.5\times10^{-4}$ | | Russell et al. (1992) | Q | 279 |
| | $5.8\times10^{-4}$ | | Suzuki et al. (1992) | Q | 232 |
| | $1.9\times10^{-4}$ | | Nirmalakhandan and Speece (1988) | Q | |
| | $5.9\times10^{-4}$ | | Irmann (1965) | Q | |
| | | 2200 | Kühne et al. (2005) | ? | |
| | $7.1\times10^{-4}$ | | Yaws (1999) | ? | 21, 80 |
| | $4.7\times10^{-4}$ | | Abraham and Weathersby (1994) | ? | 21 |
| | $7.0\times10^{-4}$ | | Yaws and Yang (1992) | ? | 21, 80 |
| difluoromethane | $6.4\times10^{-4}$ | 2100 | Kutsuna (2017) | M | 1 |
| $CH_2F_2$ | $6.8\times10^{-4}$ | 2500 | Anderson (2011) | M | |
| (R32) | $3.0\times10^{-4}$ | 3500 | Miguel et al. (2000) | M | |
| [75-10-5] | $6.9\times10^{-4}$ | 2400 | Maaßen (1995) | M | 598 |
| RWRIWBAIICGTTQ-UHFFFAOYSA-N | $6.9\times10^{-4}$ | 2300 | Reichl (1995) | M | 599 |
| | $7.9\times10^{-4}$ | | Hayer et al. (2022) | Q | 20 |
| | $1.8\times10^{-4}$ | | Gharagheizi et al. (2012) | Q | |
| | $8.4\times10^{-4}$ | | Hilal et al. (2008) | Q | |
| | | 2200 | Kühne et al. (2005) | Q | |
| | $3.1\times10^{-2}$ | | Yaffe et al. (2003) | Q | 248, 249 |
| | | 2400 | Kühne et al. (2005) | ? | |





Table A5.1: Organic fluorine (...continued)

| Substance Formula (Trivial Name) [CAS Registry Number] InChIKey | $H_s^{cp}$ (at $T^{\ominus}$) $\left[\dfrac{\mathrm{mol}}{\mathrm{m^3\,Pa}}\right]$ | $\dfrac{\mathrm{d}\ln H_s^{cp}}{\mathrm{d}(1/T)}$ [K] | Reference | Type | Note |
|---|---|---|---|---|---|
| | $8.6\times10^{-4}$ | | Yaws (1999) | ? | 21 |
| | $8.6\times10^{-4}$ | | Yaws and Yang (1992) | ? | 21 |
| trifluoromethane | $1.3\times10^{-4}$ | 2500 | Burkholder et al. (2019) | L | |
| CHF$_3$ | $1.2\times10^{-4}$ | 2200 | Burkholder et al. (2019) | L | 70 |
| (R23) | $1.3\times10^{-4}$ | 2500 | Burkholder et al. (2015) | L | |
| [75-46-7] | $1.2\times10^{-4}$ | 2200 | Burkholder et al. (2015) | L | 70 |
| XPDWGBQVDMORPB-UHFFFAOYSA-N | $1.3\times10^{-4}$ | 3300 | Sander et al. (2011) | L | |
| | $1.3\times10^{-4}$ | 3200 | Wilhelm et al. (1977) | L | |
| | $2.1\times10^{-4}$ | 2500 | Miguel et al. (2000) | M | |
| | $1.3\times10^{-4}$ | 2400 | Zheng et al. (1997) | M | 600 |
| | $1.2\times10^{-4}$ | 2400 | Maaßen (1995) | M | 601 |
| | $1.0\times10^{-4}$ | | Hine and Mookerjee (1975) | V | |
| | $1.3\times10^{-4}$ | | Yaws (2003) | X | 237 |
| | $1.0\times10^{-4}$ | | Irmann (1965) | C | |
| | $1.3\times10^{-4}$ | | Hayer et al. (2022) | Q | 20 |
| | $1.0\times10^{-4}$ | | Keshavarz et al. (2022) | Q | |
| | $8.8\times10^{-4}$ | | Duchowicz et al. (2020) | Q | 184 |
| | $1.3\times10^{-4}$ | 2700 | Li et al. (2019) | Q | 1 |
| | $4.0\times10^{-5}$ | | Gharagheizi et al. (2012) | Q | |
| | $1.6\times10^{-4}$ | | Raventos-Duran et al. (2010) | Q | 242, 243 |
| | $2.0\times10^{-4}$ | | Raventos-Duran et al. (2010) | Q | 244 |
| | $1.6\times10^{-5}$ | | Raventos-Duran et al. (2010) | Q | 245 |
| | $1.3\times10^{-4}$ | | Gharagheizi et al. (2010) | Q | 246 |
| | $2.0\times10^{-4}$ | | Hilal et al. (2008) | Q | |
| | $4.0\times10^{-5}$ | | Modarresi et al. (2007) | Q | 67 |
| | | 2200 | Kühne et al. (2005) | Q | |
| | $1.2\times10^{-4}$ | | Nirmalakhandan and Speece (1988) | Q | |
| | $1.1\times10^{-4}$ | | Irmann (1965) | Q | |
| | $1.0\times10^{-4}$ | | Duchowicz et al. (2020) | ? | 185, 21 |
| | | 3000 | Kühne et al. (2005) | ? | |
| | $1.3\times10^{-4}$ | | Yaws (1999) | ? | 21 |
| | $1.3\times10^{-4}$ | | Yaws and Yang (1992) | ? | 21 |
| tetrafluoromethane | $2.1\times10^{-6}$ | 1800 | Burkholder et al. (2019) | L | 1 |
| CF$_4$ | $1.7\times10^{-6}$ | 2300 | Burkholder et al. (2019) | L | 70 |
| (carbontetrafluoride) | $2.1\times10^{-6}$ | 1800 | Burkholder et al. (2015) | L | 1 |
| [75-73-0] | $1.7\times10^{-6}$ | 2300 | Burkholder et al. (2015) | L | 70 |
| TXEYQDLBPFQVAA-UHFFFAOYSA-N | $2.1\times10^{-6}$ | 2300 | Warneck and Williams (2012) | L | |
| | $2.1\times10^{-6}$ | 1800 | Sander et al. (2011) | L | 1 |
| | $2.1\times10^{-6}$ | 1800 | Wilhelm et al. (1977) | L | |
| | $2.0\times10^{-6}$ | 2000 | Reichl (1995) | M | 602 |
| | $2.1\times10^{-6}$ | 1800 | Scharlin and Battino (1995) | M | 603 |
| | $2.1\times10^{-6}$ | 1800 | Scharlin and Battino (1994) | M | 604 |
| | $2.1\times10^{-6}$ | | Park et al. (1982) | M | |
| | $2.1\times10^{-6}$ | 1600 | Cosgrove and Walkley (1981) | M | 11 |
| | $2.0\times10^{-6}$ | 1900 | Wen and Muccitelli (1979) | M | 605 |
| | $2.1\times10^{-6}$ | 1800 | Ashton et al. (1968) | M | 606 |



Table A5.1: Organic fluorine (...continued)

| Substance Formula (Trivial Name) [CAS Registry Number] InChIKey | $H_s^{cp}$ (at $T^{\ominus}$) $\left[\dfrac{\mathrm{mol}}{\mathrm{m^3\,Pa}}\right]$ | $\dfrac{\mathrm{d\ln} H_s^{cp}}{\mathrm{d}(1/T)}$ [K] | Reference | Type | Note |
|---|---|---|---|---|---|
| | $2.0\times10^{-6}$ | 1500 | Morrison and Johnstone (1954) | M | 607 |
| | $1.9\times10^{-6}$ | | Hine and Mookerjee (1975) | V | |
| | $3.3\times10^{-6}$ | | Pierotti (1965) | T | |
| | $1.8\times10^{-6}$ | | Yaws (2003) | X | 237 |
| | $1.9\times10^{-6}$ | | Irmann (1965) | C | |
| | $2.8\times10^{-6}$ | | Hayer et al. (2022) | Q | 20 |
| | $6.9\times10^{-7}$ | | Keshavarz et al. (2022) | Q | |
| | $1.4\times10^{-5}$ | | Duchowicz et al. (2020) | Q | |
| | $2.1\times10^{-6}$ | 1800 | Li et al. (2019) | Q | 1 |
| | $5.5\times10^{-6}$ | | Gharagheizi et al. (2012) | Q | |
| | $2.1\times10^{-6}$ | | Gharagheizi et al. (2010) | Q | 246 |
| | $9.2\times10^{-6}$ | | Hilal et al. (2008) | Q | |
| | $3.2\times10^{-6}$ | | Modarresi et al. (2007) | Q | 67 |
| | | 2200 | Kühne et al. (2005) | Q | |
| | $2.9\times10^{-6}$ | | Goss (2005) | Q | |
| | $2.0\times10^{-6}$ | | Yaffe et al. (2003) | Q | 248, 249 |
| | $1.0\times10^{-6}$ | -840 | Bonifácio et al. (2001) | Q | |
| | $1.2\times10^{-8}$ | | Katritzky et al. (1998) | Q | |
| | $5.4\times10^{-6}$ | | Nirmalakhandan and Speece (1988) | Q | |
| | $1.6\times10^{-6}$ | | Irmann (1965) | Q | |
| | $1.9\times10^{-6}$ | | Duchowicz et al. (2020) | ? | 185, 21 |
| | | 1900 | Kühne et al. (2005) | ? | |
| | $1.9\times10^{-6}$ | | Yaws (1999) | ? | 21 |
| | $2.1\times10^{-6}$ | 1700 | Yaws et al. (1999) | ? | 21 |
| | $1.8\times10^{-6}$ | | Yaws and Yang (1992) | ? | 21 |
| fluoroethane C$_2$H$_5$F [353-36-6] UHCBBWUQDAVSMS-UHFFFAOYSA-N | $4.6\times10^{-4}$ | | Yaws (2003) | X | 237 |
| | $1.6\times10^{-4}$ | | Gharagheizi et al. (2012) | Q | |
| | $3.1\times10^{-4}$ | | Raventos-Duran et al. (2010) | Q | 271, 243 |
| | $6.2\times10^{-4}$ | | Raventos-Duran et al. (2010) | Q | 244 |
| | $4.9\times10^{-4}$ | | Raventos-Duran et al. (2010) | Q | 245 |
| | $4.9\times10^{-4}$ | | Gharagheizi et al. (2010) | Q | 246 |
| | $4.8\times10^{-4}$ | | Hilal et al. (2008) | Q | |
| | $3.2\times10^{-4}$ | | Modarresi et al. (2007) | Q | 67 |
| | $4.6\times10^{-4}$ | | Yaws (1999) | ? | 21 |
| | $5.0\times10^{-4}$ | | Abraham and Weathersby (1994) | ? | 21 |
| | $4.4\times10^{-4}$ | | Yaws and Yang (1992) | ? | 21 |
| 1,1-difluoroethane C$_2$H$_4$F$_2$ (R152a) [75-37-6] NPNPZTNLOVBDOC-UHFFFAOYSA-N | $4.9\times10^{-4}$ | 2600 | Burkholder et al. (2019) | L | 608, 70 |
| | $4.9\times10^{-4}$ | 2600 | Burkholder et al. (2015) | L | 609, 70 |
| | $4.9\times10^{-4}$ | 2800 | Zheng et al. (1997) | M | 610 |
| | $5.0\times10^{-4}$ | 2800 | Maaßen (1995) | M | 611 |
| | $4.9\times10^{-4}$ | 2700 | Reichl (1995) | M | 612 |
| | $4.2\times10^{-4}$ | 2300 | McLinden (1989) | V | |
| | $4.8\times10^{-4}$ | | Hine and Mookerjee (1975) | V | |
| | $4.8\times10^{-4}$ | | Irmann (1965) | C | 294 |
| | $5.7\times10^{-4}$ | | Hayer et al. (2022) | Q | 20 |
| | $1.2\times10^{-2}$ | | Keshavarz et al. (2022) | Q | |





Table A5.1: Organic fluorine (...continued)

| Substance Formula (Trivial Name) [CAS Registry Number] InChIKey | $H_s^{cp}$ (at $T^\ominus$) $\left[\dfrac{\mathrm{mol}}{\mathrm{m^3\,Pa}}\right]$ | $\dfrac{\mathrm{d}\ln H_s^{cp}}{\mathrm{d}(1/T)}$ [K] | Reference | Type | Note |
|---|---|---|---|---|---|
| | $1.2\times10^{-3}$ | | Duchowicz et al. (2020) | Q | |
| | $1.6\times10^{-4}$ | | Gharagheizi et al. (2012) | Q | |
| | $4.9\times10^{-4}$ | | Raventos-Duran et al. (2010) | Q | 271, 243 |
| | $1.6\times10^{-4}$ | | Raventos-Duran et al. (2010) | Q | 244 |
| | $2.5\times10^{-5}$ | | Raventos-Duran et al. (2010) | Q | 245 |
| | $2.9\times10^{-4}$ | | Hilal et al. (2008) | Q | |
| | $1.4\times10^{-4}$ | | Modarresi et al. (2007) | Q | 67 |
| | | 2600 | Kühne et al. (2005) | Q | |
| | $9.0\times10^{-4}$ | | English and Carroll (2001) | Q | 230, 231 |
| | $1.4\times10^{-4}$ | | Nirmalakhandan and Speece (1988) | Q | |
| | $4.3\times10^{-4}$ | | Irmann (1965) | Q | |
| | $4.9\times10^{-4}$ | | Duchowicz et al. (2020) | ? | 185, 21 |
| | | 2800 | Kühne et al. (2005) | ? | |
| | $3.9\times10^{-4}$ | | Yaws (1999) | ? | 21, 297 |
| | $3.7\times10^{-4}$ | | Yaws and Yang (1992) | ? | 21, 297 |
| 1,2-difluoroethane C$_2$H$_4$F$_2$ [624-72-6] AHFMSNDOYCFEPH-UHFFFAOYSA-N | $2.5\times10^{-5}$ | | HSDB (2015) | Q | 99 |
| 1,1,1,2-tetrafluoroethane C$_2$H$_2$F$_4$ (R134a) [811-97-2] LVGUZGTVOIAKKC-UHFFFAOYSA-N | $1.6\times10^{-4}$ | 2700 | Burkholder et al. (2019) | L | 70 |
| | $1.6\times10^{-4}$ | 2700 | Burkholder et al. (2015) | L | 70 |
| | $1.5\times10^{-4}$ | 3100 | Ooki and Yokouchi (2011) | M | 70 |
| | $1.6\times10^{-4}$ | 2900 | Zheng et al. (1997) | M | 613 |
| | $1.6\times10^{-4}$ | 3000 | Maaßen (1995) | M | 614 |
| | $1.6\times10^{-4}$ | 2900 | Reichl (1995) | M | 615 |
| | $1.9\times10^{-4}$ | 1400 | Chang and Criddle (1995) | M | 616 |
| | $1.4\times10^{-4}$ | 2600 | McLinden (1989) | V | |
| | $2.5\times10^{-4}$ | | Hayer et al. (2022) | Q | 20 |
| | $1.5\times10^{-4}$ | 3100 | Li et al. (2019) | Q | 1 |
| | $6.5\times10^{-6}$ | | HSDB (2015) | Q | 99 |
| | $2.5\times10^{-4}$ | | Raventos-Duran et al. (2010) | Q | 271, 243 |
| | $1.2\times10^{-4}$ | | Raventos-Duran et al. (2010) | Q | 244 |
| | $6.2\times10^{-6}$ | | Raventos-Duran et al. (2010) | Q | 245 |
| | $9.7\times10^{-5}$ | | Hilal et al. (2008) | Q | |
| | $5.5\times10^{-5}$ | | Modarresi et al. (2007) | Q | 67 |
| 1,1,2,2-tetrafluoroethane C$_2$H$_2$F$_4$ [359-35-3] WXGNWUVNYMJENI-UHFFFAOYSA-N | $2.9\times10^{-4}$ | | Ebert et al. (2023) | ? | 318 |



Table A5.1: Organic fluorine (...continued)

| Substance Formula (Trivial Name) [CAS Registry Number] InChIKey | $H_s^{cp}$ (at $T^{\ominus}$) $\left[\dfrac{\mathrm{mol}}{\mathrm{m}^3\,\mathrm{Pa}}\right]$ | $\dfrac{\mathrm{d}\ln H_s^{cp}}{\mathrm{d}(1/T)}$ [K] | Reference | Type | Note |
|---|---|---|---|---|---|
| pentafluoroethane | $3.1\times10^{-4}$ | 3300 | Miguel et al. (2000) | M | |
| $C_2HF_5$ | $3.5\times10^{-5}$ | 3000 | Reichl (1995) | M | 617 |
| (R125) | $8.0\times10^{-5}$ | 4800 | McLinden (1989) | V | |
| [354-33-6] | $1.3\times10^{-4}$ | | Hayer et al. (2022) | Q | 20 |
| GTLACDSXYULKMZ-UHFFFAOYSA-N | $3.5\times10^{-5}$ | 3000 | Li et al. (2019) | Q | 1 |
| | $2.0\times10^{-4}$ | | HSDB (2015) | Q | 99 |
| | $3.2\times10^{-6}$ | | Zhang et al. (2010) | Q | 287, 288 |
| | $2.0\times10^{-5}$ | | Zhang et al. (2010) | Q | 287, 289 |
| | $5.7\times10^{-5}$ | | Zhang et al. (2010) | Q | 287, 290 |
| | $2.1\times10^{-5}$ | | Zhang et al. (2010) | Q | 287, 291 |
| | | 2600 | Kühne et al. (2005) | Q | |
| | | 2900 | Kühne et al. (2005) | ? | |
| hexafluoroethane | $6.5\times10^{-7}$ | 2100 | Bonifácio et al. (2001) | M | |
| $C_2F_6$ | $5.3\times10^{-7}$ | | Park et al. (1982) | M | |
| [76-16-4] | $5.6\times10^{-7}$ | 2300 | Wen and Muccitelli (1979) | M | 618 |
| WMIYKQLTONQJES-UHFFFAOYSA-N | $5.8\times10^{-7}$ | | Yaws (2003) | X | 237 |
| | $7.1\times10^{-7}$ | | Hayer et al. (2022) | Q | 20 |
| | $9.3\times10^{-7}$ | | Keshavarz et al. (2022) | Q | |
| | $2.2\times10^{-5}$ | | Duchowicz et al. (2020) | Q | |
| | $5.8\times10^{-7}$ | 2600 | Li et al. (2019) | Q | 1 |
| | $1.5\times10^{-5}$ | | Gharagheizi et al. (2012) | Q | |
| | $4.1\times10^{-7}$ | | Zhang et al. (2010) | Q | 287, 288 |
| | $1.1\times10^{-5}$ | | Zhang et al. (2010) | Q | 287, 289 |
| | $8.4\times10^{-7}$ | | Zhang et al. (2010) | Q | 287, 290 |
| | $1.9\times10^{-6}$ | | Zhang et al. (2010) | Q | 287, 291 |
| | $8.1\times10^{-7}$ | | Gharagheizi et al. (2010) | Q | 246 |
| | $1.2\times10^{-5}$ | | Hilal et al. (2008) | Q | |
| | $1.8\times10^{-6}$ | | Modarresi et al. (2007) | Q | 67 |
| | | 2600 | Kühne et al. (2005) | Q | |
| | $1.2\times10^{-6}$ | 1700 | Bonifácio et al. (2001) | Q | |
| | $4.9\times10^{-7}$ | | Duchowicz et al. (2020) | ? | 185, 21 |
| | | 2900 | Kühne et al. (2005) | ? | |
| | $5.8\times10^{-7}$ | | Yaws (1999) | ? | 21 |
| | $5.8\times10^{-7}$ | | Yaws and Yang (1992) | ? | 21 |
| 1-fluoropropane | $6.3\times10^{-4}$ | | Yaws (2003) | X | 237, 619 |
| $C_3H_7F$ | $7.9\times10^{-5}$ | | Gharagheizi et al. (2012) | Q | |
| [460-13-9] | $5.8\times10^{-4}$ | | Gharagheizi et al. (2010) | Q | 246 |
| JRHNUZCXXOTJCA-UHFFFAOYSA-N | $5.7\times10^{-4}$ | | Hilal et al. (2008) | Q | |
| | $6.2\times10^{-4}$ | | Yaws (1999) | ? | 21, 619 |
| | $3.6\times10^{-4}$ | | Abraham and Weathersby (1994) | ? | 21 |
| | $6.1\times10^{-4}$ | | Yaws and Yang (1992) | ? | 21, 619 |



Table A5.1: Organic fluorine (...continued)

| Substance Formula (Trivial Name) [CAS Registry Number] InChIKey | $H_s^{cp}$ (at $T^\ominus$) $\left[\dfrac{\mathrm{mol}}{\mathrm{m^3\,Pa}}\right]$ | $\dfrac{\mathrm{d\ln} H_s^{cp}}{\mathrm{d}(1/T)}$ [K] | Reference | Type | Note |
|---|---|---|---|---|---|
| 2-fluoropropane | $6.0\times10^{-4}$ | | Yaws (2003) | X | 237, 80 |
| $C_3H_7F$ | $2.2\times10^{-4}$ | | Gharagheizi et al. (2012) | Q | |
| [420-26-8] | $6.0\times10^{-4}$ | | Gharagheizi et al. (2010) | Q | 246 |
| PRNZBCYBKGCOFI-UHFFFAOYSA-N | $2.5\times10^{-4}$ | | Hilal et al. (2008) | Q | |
| | $5.9\times10^{-4}$ | | Yaws (1999) | ? | 21, 80 |
| | $3.8\times10^{-4}$ | | Abraham and Weathersby (1994) | ? | 21 |
| | $5.8\times10^{-4}$ | | Yaws and Yang (1992) | ? | 21, 80 |
| 1,1,1,2,2-pentafluoropropane | $3.0\times10^{-1}$ | | Nirmalakhandan and Speece (1988) | Q | |
| $C_3H_3F_5$ | | | | | |
| [1814-88-6] | | | | | |
| FDOPVENYMZRARC-UHFFFAOYSA-N | | | | | |
| 1,1,1,3,3,3-hexafluoropropane | $1.2\times10^{-6}$ | | Zhang et al. (2010) | Q | 287, 288 |
| $C_3H_2F_6$ | $3.9\times10^{-5}$ | | Zhang et al. (2010) | Q | 287, 289 |
| [690-39-1] | $1.8\times10^{-4}$ | | Zhang et al. (2010) | Q | 287, 290 |
| NSGXIBWMJZWTPY-UHFFFAOYSA-N | $2.7\times10^{-6}$ | | Zhang et al. (2010) | Q | 287, 291 |
| 1,1,1,2,3,3,3-heptafluoropropane | $1.4\times10^{-5}$ | 3300 | Reichl (1995) | M | 620 |
| $C_3HF_7$ | $2.2\times10^{-4}$ | | Hayer et al. (2022) | Q | 20 |
| (R227) | $6.2\times10^{-7}$ | | HSDB (2015) | Q | 99 |
| [431-89-0] | | 2900 | Kühne et al. (2005) | Q | |
| YFMFNYKEUDLDTL-UHFFFAOYSA-N | | 3300 | Kühne et al. (2005) | ? | |
| octafluoropropane | $1.2\times10^{-7}$ | 6900 | Wen and Muccitelli (1979) | M | |
| $C_3F_8$ | $3.0\times10^{-7}$ | | Duchowicz et al. (2020) | V | 186 |
| (R218) | $3.0\times10^{-7}$ | | HSDB (2015) | V | |
| [76-19-7] | $3.2\times10^{-7}$ | | Yaws (2003) | X | 237, 80 |
| QYSGYZVSCZSLHT-UHFFFAOYSA-N | $1.1\times10^{-3}$ | | Hayer et al. (2022) | Q | 20 |
| | $5.1\times10^{-5}$ | | Duchowicz et al. (2020) | Q | |
| | $4.2\times10^{-5}$ | | Gharagheizi et al. (2012) | Q | |
| | $7.7\times10^{-8}$ | | Zhang et al. (2010) | Q | 287, 288 |
| | $1.0\times10^{-5}$ | | Zhang et al. (2010) | Q | 287, 289 |
| | $3.8\times10^{-7}$ | | Zhang et al. (2010) | Q | 287, 290 |
| | $4.5\times10^{-7}$ | | Zhang et al. (2010) | Q | 287, 291 |
| | $3.0\times10^{-7}$ | | Gharagheizi et al. (2010) | Q | 246 |
| | $1.1\times10^{-5}$ | | Hilal et al. (2008) | Q | |
| | $3.1\times10^{-7}$ | | Yaws (1999) | ? | 21, 80 |
| decafluorobutane | $1.5\times10^{-8}$ | | HSDB (2015) | Q | 99 |
| $C_4F_{10}$ | | | | | |
| [355-25-9] | | | | | |
| KAVGMUDTWQVPDF-UHFFFAOYSA-N | | | | | |





Table A5.1: Organic fluorine (. . . continued)

| Substance Formula (Trivial Name) [CAS Registry Number] InChIKey | $H_s^{cp}$ (at $T^\ominus$) $\left[\dfrac{\mathrm{mol}}{\mathrm{m^3\,Pa}}\right]$ | $\dfrac{\mathrm{d}\ln H_s^{cp}}{\mathrm{d}(1/T)}$ [K] | Reference | Type | Note |
|---|---|---|---|---|---|
| octafluorocyclobutane | $1.2\times10^{-6}$ | 3000 | Clever et al. (2005) | L | 621, 622 |
| $C_4F_8$ | $1.3\times10^{-6}$ | 3300 | Scharlin and Battino (1994) | M | 623 |
| [115-25-3] | $1.2\times10^{-6}$ | | Park et al. (1982) | M | |
| BCCOBQSFUDVTJQ-UHFFFAOYSA-N | $1.2\times10^{-6}$ | 3000 | Wen and Muccitelli (1979) | M | 624 |
| | $1.2\times10^{-6}$ | | Duchowicz et al. (2020) | V | 186 |
| | $2.5\times10^{-6}$ | | Yaws (2003) | X | 237, 372 |
| | $1.6\times10^{-6}$ | | Hayer et al. (2022) | Q | 20 |
| | $3.9\times10^{-3}$ | | Duchowicz et al. (2020) | Q | |
| | $1.0\times10^{-4}$ | | Gharagheizi et al. (2012) | Q | |
| | $1.3\times10^{-7}$ | | Zhang et al. (2010) | Q | 287, 288 |
| | $1.6\times10^{-6}$ | | Zhang et al. (2010) | Q | 287, 289 |
| | $2.2\times10^{-6}$ | | Zhang et al. (2010) | Q | 287, 290 |
| | $1.0\times10^{-6}$ | | Zhang et al. (2010) | Q | 287, 291 |
| | $2.6\times10^{-6}$ | | Gharagheizi et al. (2010) | Q | 246 |
| | $9.2\times10^{-6}$ | | Hilal et al. (2008) | Q | |
| | | 4500 | Kühne et al. (2005) | Q | |
| | | 3800 | Kühne et al. (2005) | ? | |
| | $2.6\times10^{-6}$ | | Yaws (1999) | ? | 21, 372 |
| | $2.5\times10^{-6}$ | | Yaws and Yang (1992) | ? | 21, 372 |
| 1,1,1,2,2,3,4,5,5,5-decafluoropentane | $4.4\times10^{-8}$ | | Zhang et al. (2010) | Q | 287, 288 |
| $C_5H_2F_{10}$ | $3.2\times10^{-5}$ | | Zhang et al. (2010) | Q | 287, 289 |
| [138495-42-8] | $1.8\times10^{-4}$ | | Zhang et al. (2010) | Q | 287, 290 |
| RIQRGMUSBYGDBL-UHFFFAOYSA-N | $9.0\times10^{-7}$ | | Zhang et al. (2010) | Q | 287, 291 |
| dodecafluoropentane | $4.6\times10^{-8}$ | | Brockbank (2013) | L | |
| $C_5F_{12}$ | $6.1\times10^{-6}$ | | Hilal et al. (2008) | Q | |
| [678-26-2] | | | | | |
| NJCBUSHGCBERSK-UHFFFAOYSA-N | | | | | |
| tetradecafluorohexane | $9.3\times10^{-9}$ | | Brockbank (2013) | L | |
| $C_6F_{14}$ | $5.4\times10^{-10}$ | | HSDB (2015) | Q | 99 |
| (perflexane) | | | | | |
| [355-42-0] | | | | | |
| ZJIJAJXFLBMLCK-UHFFFAOYSA-N | | | | | |
| fluorocyclohexane | $1.3\times10^{-3}$ | | Hilal et al. (2008) | Q | |
| $C_6H_{11}F$ | | | | | |
| [372-46-3] | | | | | |
| GOBGVVAHHOUMDK-UHFFFAOYSA-N | | | | | |
| 1-fluoroheptane | $2.7\times10^{-4}$ | | Hilal et al. (2008) | Q | |
| $C_7H_{15}F$ | | | | | |
| [661-11-0] | | | | | |
| BITLXSQYFZTQGC-UHFFFAOYSA-N | | | | | |



Table A5.1: Organic fluorine (. . . continued)

| Substance Formula (Trivial Name) [CAS Registry Number] InChIKey | $H_s^{cp}$ (at $T^{\ominus}$) $\left[\dfrac{\mathrm{mol}}{\mathrm{m^3\,Pa}}\right]$ | $\dfrac{\mathrm{d}\ln H_s^{cp}}{\mathrm{d}(1/T)}$ [K] | Reference | Type | Note |
|---|---|---|---|---|---|
| hexadecafluoroheptane C$_7$F$_{16}$ [335-57-9] LGUZHRODIJCVOC-UHFFFAOYSA-N | $3.0\times10^{-9}$ $1.9\times10^{-7}$ | | Brockbank (2013) Hilal et al. (2008) | L Q | |
| 1-fluorooctane C$_8$H$_{17}$F [463-11-6] DHIVLKMGKIZOHF-UHFFFAOYSA-N | $1.5\times10^{-4}$ | | Hilal et al. (2008) | Q | |
| perfluorooctane C$_8$F$_{18}$ [307-34-6] YVBBRRALBYAZBM-UHFFFAOYSA-N | $8.0\times10^{-10}$ | | Brockbank (2013) | L | |
| eicosafluorononane C$_9$F$_{20}$ [375-96-2] UVWPNDVAQBNQBG-UHFFFAOYSA-N | $4.5\times10^{-9}$ | | Hilal et al. (2008) | Q | |
| perfluoroundecane C$_{11}$F$_{24}$ [307-49-3] VCIVYCHKSHULON-UHFFFAOYSA-N | $1.3\times10^{-13}$ $1.2\times10^{-11}$ $1.2\times10^{-9}$ $6.0\times10^{-12}$ | | Zhang et al. (2010) Zhang et al. (2010) Zhang et al. (2010) Zhang et al. (2010) | Q Q Q Q | 287, 288 287, 289 287, 290 287, 291 |
| 1,1,1,2,2,3,3,4,4,5,5,6,6,7,7,8,8,9,9,10,10-henicosafluorododecane C$_{12}$H$_5$F$_{21}$ (perfluorodecyl ethane) [154478-87-2] HUPGRQWHZOWFPQ-UHFFFAOYSA-N | $5.1\times10^{-10}$ | | Plassmann et al. (2010) | Q | |
| 1,1,1,2,2,3,3,4,4,5,5,6,6-tridecafluorotetradecane C$_{14}$H$_{17}$F$_{13}$ [133331-77-8] WRYIIOKOQSICTB-UHFFFAOYSA-N | $6.4\times10^{-7}$ | | Plassmann et al. (2010) | Q | |
| 1,1,1,2,2,3,3,4,4,5,5,6,6-tridecafluoroicosane C$_{20}$H$_{29}$F$_{13}$ [154628-00-9] BREOHRVZEZMFOB-UHFFFAOYSA-N | $2.5\times10^{-7}$ | | Plassmann et al. (2010) | Q | |
| 1,1,1,2,2,3,3,4,4,5,5,6,6-tridecafluorodocosane C$_{22}$H$_{33}$F$_{13}$ [133310-71-1] ZKYMFADZZFTYJH-UHFFFAOYSA-N | $2.0\times10^{-7}$ | | Plassmann et al. (2010) | Q | |



Table A5.1: Organic fluorine (. . . continued)

| Substance Formula (Trivial Name) [CAS Registry Number] InChIKey | $H_s^{cp}$ (at $T^{\ominus}$) $\left[\dfrac{\text{mol}}{\text{m}^3\,\text{Pa}}\right]$ | $\dfrac{\mathrm{d}\ln H_s^{cp}}{\mathrm{d}(1/T)}$ [K] | Reference | Type | Note |
|---|---|---|---|---|---|
| 1,1,1,2,2,3,3,4,4,5,5,6,6,7,7,8,8-heptadecafluorotetracosane $C_{24}H_{33}F_{17}$ [117146-18-6] FTECWCULPOFDJS-UHFFFAOYSA-N | $4.0\times10^{-9}$ | | Plassmann et al. (2010) | Q | |
| 1,1,1,2,2,3,3,4,4,5,5,6,6,7,7,8,8,9,9,10,10-henicosafluorohexacosane $C_{26}H_{33}F_{21}$ LZENXKBSJNMIKY-UHFFFAOYSA-N | $3.2\times10^{-11}$ | | Plassmann et al. (2010) | Q | |
| 1,1,1,2,2,3,3,4,4,5,5,6,6,7,7,8,8,9,9,10,10,11,11,12,12-pentacosafluorohexacosane $C_{26}H_{29}F_{25}$ [93454-73-0] OUASUHCMZXPMCH-UHFFFAOYSA-N | $1.6\times10^{-13}$ | | Plassmann et al. (2010) | Q | |
| 1,1,1,2,2,3,3,4,4,5,5,6,6,7,7,8,8,9,9,10,10,11,11,12,12-pentacosafluorooctacosane $C_{28}H_{33}F_{25}$ [93454-74-1] CZCMNCOQMXNFTL-UHFFFAOYSA-N | $8.0\times10^{-14}$ | | Plassmann et al. (2010) | Q | |
| fluoroethene $C_2H_3F$ (vinyl fluoride) [75-02-5] XUCNUKMRBVNAPB-UHFFFAOYSA-N | $8.2\times10^{-5}$ | | HSDB (2015) | Q | 99 |
| 1,1-difluoroethene $C_2H_2F_2$ [75-38-7] BQCIDUSAKPWEOX-UHFFFAOYSA-N | $2.5\times10^{-5}$ $2.6\times10^{-5}$ $2.8\times10^{-5}$ $2.8\times10^{-5}$ $5.1\times10^{-5}$ $2.9\times10^{-5}$ $1.6\times10^{-5}$ $2.6\times10^{-5}$ $2.5\times10^{-5}$ | | HSDB (2015) Yaws (2003) Gharagheizi et al. (2012) Gharagheizi et al. (2010) Hilal et al. (2008) Yaffe et al. (2003) Katritzky et al. (1998) Yaws (1999) Yaws and Yang (1992) | V X Q Q Q Q Q ? ? | 237 246 248, 249 21 21 |
| trifluoroethene $C_2HF_3$ [359-11-5] MIZLGWKEZAPEFJ-UHFFFAOYSA-N | $2.3\times10^{-5}$ | | HSDB (2015) | Q | 99 |





Table A5.1: Organic fluorine (. . . continued)

| Substance Formula (Trivial Name) [CAS Registry Number] InChIKey | $H_s^{cp}$ (at $T^{\ominus}$) $\left[\dfrac{\text{mol}}{\text{m}^3\,\text{Pa}}\right]$ | $\dfrac{\mathrm{d}\ln H_s^{cp}}{\mathrm{d}(1/T)}$ [K] | Reference | Type | Note |
|---|---|---|---|---|---|
| tetrafluoroethene $C_2F_4$ [116-14-3] BFKJFAAPBSQJPD-UHFFFAOYSA-N | $1.6\times10^{-5}$ | 2100 | Wilhelm et al. (1977) | L | |
| | $1.6\times10^{-5}$ | | HSDB (2015) | V | |
| | $1.6\times10^{-5}$ | | Yaws (2003) | X | 237 |
| | $9.8\times10^{-6}$ | | Irmann (1965) | C | 38 |
| | $1.4\times10^{-5}$ | | Hayer et al. (2022) | Q | 20 |
| | $2.8\times10^{-5}$ | | Gharagheizi et al. (2012) | Q | |
| | $1.6\times10^{-5}$ | | Gharagheizi et al. (2010) | Q | 246 |
| | $1.9\times10^{-5}$ | | Hilal et al. (2008) | Q | |
| | | 2400 | Kühne et al. (2005) | Q | |
| | $2.8\times10^{-5}$ | | Yao et al. (2002) | Q | 229 |
| | | 2100 | Kühne et al. (2005) | ? | |
| | $1.6\times10^{-5}$ | | Yaws (1999) | ? | 21 |
| | $1.6\times10^{-5}$ | | Yaws and Yang (1992) | ? | 21 |
| 3,3,3-trifluoropropene $C_3H_3F_3$ [677-21-4] FDMFUZHCIRHGRG-UHFFFAOYSA-N | $1.3\times10^{-5}$ | | HSDB (2015) | Q | 99 |
| hexafluoropropene $C_3F_6$ [116-15-4] HCDGVLDPFQMKDK-UHFFFAOYSA-N | $2.9\times10^{-6}$ | 2400 | Wilhelm et al. (1977) | L | |
| | $6.8\times10^{-6}$ | 2600 | Maaßen (1995) | M | 625 |
| | $7.3\times10^{-6}$ | | Hayer et al. (2022) | Q | 20 |
| | $1.8\times10^{-6}$ | | HSDB (2015) | Q | 99 |
| | $3.6\times10^{-5}$ | | Hilal et al. (2008) | Q | |
| | | 2800 | Kühne et al. (2005) | Q | |
| | | 2400 | Kühne et al. (2005) | ? | |
| 1,1,3,3-pentafluoro-2-(trifluoromethyl)-1-propene $C_4F_8$ (perfluoroisobutylene) [382-21-8] DAFIBNSJXIGBQB-UHFFFAOYSA-N | $2.9\times10^{-7}$ | | HSDB (2015) | Q | 99 |
| (perfluorobutyl)ethene $C_6H_3F_9$ (4:2 FTO) [19430-93-4] GVEUEBXMTMZVSD-UHFFFAOYSA-N | $9.0\times10^{-8}$ | | HSDB (2015) | Q | 99 |
| | $8.8\times10^{-8}$ | | Zhang et al. (2010) | Q | 287, 288 |
| | $3.3\times10^{-6}$ | | Zhang et al. (2010) | Q | 287, 289 |
| | $8.6\times10^{-6}$ | | Zhang et al. (2010) | Q | 287, 290 |
| | $3.6\times10^{-7}$ | | Zhang et al. (2010) | Q | 287, 291 |
| | $2.5\times10^{-6}$ | 4100 | Goss et al. (2006) | Q | |
| (E)-perfluoro(4-methyl-2-pentene) $C_6F_{12}$ [3709-71-5] SAPOZTRFWJZUFT-OWOJBTEDSA-N | $6.4\times10^{-8}$ | | Ebert et al. (2023) | ? | 365 |





Table A5.1: Organic fluorine (...continued)

| Substance Formula (Trivial Name) [CAS Registry Number] InChIKey | $H_s^{cp}$ (at $T^\ominus$) $\left[\dfrac{\text{mol}}{\text{m}^3\,\text{Pa}}\right]$ | $\dfrac{\text{d}\ln H_s^{cp}}{\text{d}(1/T)}$ [K] | Reference | Type | Note |
|---|---|---|---|---|---|
| (perfluorohexyl)ethene $C_8H_3F_{13}$ (6:2 FTO) [25291-17-2] FYQFWFHDPNXORA-UHFFFAOYSA-N | $1.1\times10^{-4}$ $5.3\times10^{-7}$ | 4900 4900 | Abusallout et al. (2022) Goss et al. (2006) | M Q | |
| (perfluorooctyl)ethene $C_{10}H_3F_{17}$ (8:2 FTO) [21652-58-4] NKAMGQZDVMQEJL-UHFFFAOYSA-N | $1.4\times10^{-7}$ | 5700 | Goss et al. (2006) | Q | |
| (perfluorodecyl)ethene $C_{12}H_3F_{21}$ (10:2 FTO) [30389-25-4] UCHSAVGOZUCXHC-UHFFFAOYSA-N | $3.3\times10^{-8}$ | 6500 | Goss et al. (2006) | Q | |
| 1,1,1,2,2,3,3,4,4,5,5,6,6,7,7,8,8,9,9, 10,10,11,11,12,12- pentacosafluorooctacos-13-ene $C_{28}H_{31}F_{25}$ LBVDHZOJOFOXJM-FOCLMDBBSA-N | $8.0\times10^{-11}$ | | Plassmann et al. (2010) | Q | |
| fluorobenzene $C_6H_5F$ [462-06-6] PYLWMHQQBFSUBP-UHFFFAOYSA-N | $1.5\times10^{-3}$ $1.6\times10^{-3}$ $1.6\times10^{-3}$ $1.4\times10^{-3}$ $1.1\times10^{-3}$ $1.5\times10^{-3}$ $1.6\times10^{-3}$ $1.4\times10^{-3}$ $1.6\times10^{-3}$ $1.6\times10^{-3}$ $7.4\times10^{-3}$ $8.6\times10^{-3}$ $1.6\times10^{-3}$ $2.2\times10^{-3}$ $4.1\times10^{-3}$ $2.5\times10^{-3}$ $1.6\times10^{-3}$ $1.6\times10^{-3}$ $1.6\times10^{-3}$ $2.0\times10^{-3}$ $1.4\times10^{-3}$ $1.6\times10^{-3}$ $2.3\times10^{-3}$ $1.3\times10^{-3}$ | 4100 3900 4300 4400 3700 | Brockbank (2013) Mackay and Shiu (1981) Hiatt (2013) Dewulf et al. (1999) Li and Carr (1993) Hartkopf and Karger (1973) Schüürmann (2000) Mackay et al. (1993) Yaws (2003) Sieg et al. (2008) Keshavarz et al. (2022) Duchowicz et al. (2020) Li et al. (2014) Li et al. (2014) Gharagheizi et al. (2012) Raventos-Duran et al. (2010) Raventos-Duran et al. (2010) Raventos-Duran et al. (2010) Gharagheizi et al. (2010) Hilal et al. (2008) Modarresi et al. (2007) Kühne et al. (2005) Yaffe et al. (2003) Yao et al. (2002) Katritzky et al. (1998) | L L M M M M V V X C Q Q Q Q Q Q Q Q Q Q Q Q Q Q Q | 1 237 299 241 241 242, 243 244 245 246 67 248, 249 229, 267 |



Table A5.1: Organic fluorine (...continued)

| Substance Formula (Trivial Name) [CAS Registry Number] InChIKey | $H_s^{cp}$ (at $T^\ominus$) $\left[\dfrac{\mathrm{mol}}{\mathrm{m^3\,Pa}}\right]$ | $\dfrac{\mathrm{d}\ln H_s^{cp}}{\mathrm{d}(1/T)}$ [K] | Reference | Type | Note |
|---|---|---|---|---|---|
| | $5.0\times10^{-3}$ | | Nirmalakhandan et al. (1997) | Q | |
| | $1.6\times10^{-3}$ | | Duchowicz et al. (2020) | ? | 185, 21 |
| | | 3800 | Kühne et al. (2005) | ? | |
| | $1.6\times10^{-3}$ | | Yaws (1999) | ? | 21 |
| | $1.2\times10^{-3}$ | | Hoff et al. (1993) | ? | 21 |
| | $1.6\times10^{-3}$ | | Yaws and Yang (1992) | ? | 21 |
| | $1.5\times10^{-3}$ | | Abraham et al. (1990) | ? | |
| 1,2-difluorobenzene $C_6H_4F_2$ (*o*-difluorobenzene) [367-11-3] GOYDNIKZWGIXJT-UHFFFAOYSA-N | $1.3\times10^{-3}$ $1.2\times10^{-3}$ $1.4\times10^{-3}$ $3.6\times10^{-3}$ $1.4\times10^{-3}$ $2.2\times10^{-3}$ $6.4\times10^{-4}$ $5.1\times10^{-3}$ $1.4\times10^{-3}$ $1.4\times10^{-3}$ | 3700 3500 | Brockbank (2013) Brockbank et al. (2013) Yaws (2003) Gharagheizi et al. (2012) Gharagheizi et al. (2010) Hilal et al. (2008) Modarresi et al. (2007) Yao et al. (2002) Yaws (1999) Yaws and Yang (1992) | L M X Q Q Q Q Q ? ? | 1 237 246 67 229 21 21 |
| 1,3-difluorobenzene $C_6H_4F_2$ (*m*-difluorobenzene) [372-18-9] UEMGWPRHOOEKTA-UHFFFAOYSA-N | $3.6\times10^{-3}$ $1.3\times10^{-3}$ $5.2\times10^{-4}$ $1.3\times10^{-4}$ $1.3\times10^{-4}$ | | Gharagheizi et al. (2012) Hilal et al. (2008) Modarresi et al. (2007) Yaws (1999) Yaws and Yang (1992) | Q Q Q ? ? | 67 21 21 |
| 1,4-difluorobenzene $C_6H_4F_2$ (*p*-difluorobenzene) [540-36-3] QUGUFLJIAFISSW-UHFFFAOYSA-N | $1.6\times10^{-3}$ $1.3\times10^{-3}$ $4.2\times10^{-3}$ $1.4\times10^{-3}$ $1.8\times10^{-3}$ $3.5\times10^{-4}$ $9.5\times10^{-4}$ $1.3\times10^{-3}$ $1.3\times10^{-3}$ | 3900 | Hiatt (2013) Yaws (2003) Gharagheizi et al. (2012) Gharagheizi et al. (2010) Hilal et al. (2008) Modarresi et al. (2007) Yao et al. (2002) Yaws (1999) Yaws and Yang (1992) | M X Q Q Q Q Q ? ? | 237 246 67 229 21 21 |
| 1,2,3,5-tetrafluorobenzene $C_6H_2F_4$ [2367-82-0] UHHYOKRQTQBKSB-UHFFFAOYSA-N | $5.1\times10^{-4}$ $1.2\times10^{-2}$ $5.0\times10^{-4}$ $9.4\times10^{-5}$ | | Duchowicz et al. (2020) Duchowicz et al. (2020) Hilal et al. (2008) Modarresi et al. (2007) | V Q Q Q | 186 67 |
| 1,2,4,5-tetrafluorobenzene $C_6H_2F_4$ [327-54-8] SDXUIOOHCIQXRP-UHFFFAOYSA-N | $5.6\times10^{-4}$ $1.1\times10^{-2}$ $7.0\times10^{-4}$ $1.2\times10^{-4}$ | | Duchowicz et al. (2020) Duchowicz et al. (2020) Hilal et al. (2008) Modarresi et al. (2007) | V Q Q Q | 186 67 |
| pentafluorobenzene $C_6HF_5$ [363-72-4] WACNXHCZHTVBJM-UHFFFAOYSA-N | $7.5\times10^{-4}$ | 4800 | Hiatt (2013) | M | |



Table A5.1: Organic fluorine (...continued)

| Substance Formula (Trivial Name) [CAS Registry Number] InChIKey | $H_s^{cp}$ (at $T^{\ominus}$) $\left[\dfrac{\mathrm{mol}}{\mathrm{m^3\,Pa}}\right]$ | $\dfrac{\mathrm{d\ln}H_s^{cp}}{\mathrm{d}(1/T)}$ [K] | Reference | Type | Note |
|---|---|---|---|---|---|
| hexafluorobenzene | $3.0{\times}10^{-4}$ | 5100 | Brockbank (2013) | L | 1 |
| $C_6F_6$ | $5.5{\times}10^{-4}$ | 5200 | Hiatt (2013) | M | |
| [392-56-3] | $2.9{\times}10^{-4}$ | | Schröder et al. (2011) | M | |
| ZQBFAOFFOQMSGJ-UHFFFAOYSA-N | $1.1{\times}10^{-4}$ | | Schröder et al. (2011) | Q | 626 |
| (trifluoromethyl)-benzene | $6.0{\times}10^{-4}$ | | Duchowicz et al. (2020) | V | 186 |
| $C_6H_5CF_3$ | $5.8{\times}10^{-4}$ | | HSDB (2015) | V | |
| ($\alpha,\alpha,\alpha$-trifluorotoluene; | $6.1{\times}10^{-4}$ | | Abraham et al. (1994a) | V | |
| benzotrifluoride) | | | | | |
| [98-08-8] | $6.2{\times}10^{-4}$ | | Mackay and Shiu (1981) | V | |
| GETTZEONDQJALK-UHFFFAOYSA-N | $5.9{\times}10^{-4}$ | | Yaws (2003) | X | 237 |
| | $3.5{\times}10^{-3}$ | | Duchowicz et al. (2020) | Q | |
| | $3.2{\times}10^{-3}$ | | Gharagheizi et al. (2012) | Q | |
| | $7.8{\times}10^{-3}$ | | Raventos-Duran et al. (2010) | Q | 242, 243 |
| | $1.2{\times}10^{-3}$ | | Raventos-Duran et al. (2010) | Q | 244 |
| | $2.0{\times}10^{-4}$ | | Raventos-Duran et al. (2010) | Q | 245 |
| | $5.7{\times}10^{-4}$ | | Gharagheizi et al. (2010) | Q | 246 |
| | $1.3{\times}10^{-3}$ | | Hilal et al. (2008) | Q | |
| | $2.7{\times}10^{-4}$ | | Modarresi et al. (2007) | Q | 67 |
| | $6.2{\times}10^{-3}$ | | Yaffe et al. (2003) | Q | 248, 249 |
| | $6.1{\times}10^{-4}$ | | Yaffe et al. (2003) | Q | 248, 249 |
| | $1.9{\times}10^{-2}$ | | Nirmalakhandan et al. (1997) | Q | |
| | $6.0{\times}10^{-4}$ | | Yaws (1999) | ? | 21 |
| | $6.0{\times}10^{-4}$ | | Yaws and Yang (1992) | ? | 21 |
| decafluorobiphenyl | $6.7{\times}10^{-3}$ | 3600 | Hiatt (2013) | M | |
| $C_{10}F_{10}$ | | | | | |
| [434-90-2] | | | | | |
| ONUFSRWQCKNVSL-UHFFFAOYSA-N | | | | | |
| carbonyl fluoride | $3.5{\times}10^{-1}$ | | Mirabel et al. (1996) | M | |
| $COF_2$ | $9.9{\times}10^{-3}$ | | De Bruyn et al. (1995a) | M | 449 |
| [353-50-4] | $2.0{\times}10^{-1}$ | | George et al. (1993) | X | 627 |
| IYRWEQXVUNLMAY-UHFFFAOYSA-N | | | | | |
| formyl fluoride | $3.0{\times}10^{-2}$ | | Kanakidou et al. (1995) | E | |
| FCHO | | | | | |
| [1493-02-3] | | | | | |
| NHGVZTMBVDFPHJ-UHFFFAOYSA-N | | | | | |
| 2-fluoroethanol | $4.6{\times}10^{-1}$ | | Burkholder et al. (2019) | L | |
| $C_2H_5FO$ | $4.6{\times}10^{-1}$ | | Burkholder et al. (2015) | L | |
| [371-62-0] | $4.6{\times}10^{-1}$ | | O'Farrell and Waghorne (2010) | M | |
| GGDYAKVUZMZKRV-UHFFFAOYSA-N | 5.5 | | Duchowicz et al. (2020) | V | 186 |
| | 1.9 | | Duchowicz et al. (2020) | Q | |
| | 1.4 | | HSDB (2015) | Q | 99 |
| | 2.5 | | Hilal et al. (2008) | Q | |
| | 5.6 | | Modarresi et al. (2007) | Q | 67 |



Table A5.1: Organic fluorine (...continued)

| Substance<br>Formula<br>(Trivial Name)<br>[CAS Registry Number]<br>InChIKey | $H_s^{cp}$<br>(at $T^\ominus$)<br>$\left[\dfrac{\text{mol}}{\text{m}^3\,\text{Pa}}\right]$ | $\dfrac{\text{d}\ln H_s^{cp}}{\text{d}(1/T)}$<br><br>[K] | Reference | Type | Note |
|---|---|---|---|---|---|
| 2,2-difluoroethanol | $9.3\times10^{-1}$ | | Burkholder et al. (2019) | L | |
| $C_2H_4F_2O$ | $9.3\times10^{-1}$ | | Burkholder et al. (2015) | L | |
| [359-13-7] | $9.2\times10^{-1}$ | | O'Farrell and Waghorne (2010) | M | |
| VOGSDFLJZPNWHY-UHFFFAOYSA-N | | | | | |
| 2,2,2-trifluoroethanol | $4.7\times10^{-1}$ | 6200 | Burkholder et al. (2019) | L | |
| $CF_3CH_2OH$ | $4.7\times10^{-1}$ | 6200 | Burkholder et al. (2015) | L | |
| [75-89-8] | $4.7\times10^{-1}$ | 6200 | Sander et al. (2011) | L | |
| RHQDFWAXVIIEBN-UHFFFAOYSA-N | $4.7\times10^{-1}$ | 6200 | Chen et al. (2003) | M | |
| | $4.3\times10^{-1}$ | | Eger et al. (1999) | M | 14 |
| | $5.8\times10^{-1}$ | 5900 | Rochester and Symonds (1973) | M | |
| | $3.4\times10^{-1}$ | | Keshavarz et al. (2022) | Q | |
| | $8.6\times10^{-1}$ | | Duchowicz et al. (2020) | Q | |
| | $3.9\times10^{-1}$ | | Raventos-Duran et al. (2010) | Q | 242, 243 |
| | $2.5\times10^{-1}$ | | Raventos-Duran et al. (2010) | Q | 244 |
| | $3.1\times10^{-1}$ | | Raventos-Duran et al. (2010) | Q | 245 |
| | $3.5\times10^{-1}$ | | Zhang et al. (2010) | Q | 287, 288 |
| | $2.4\times10^{-1}$ | | Zhang et al. (2010) | Q | 287, 289 |
| | 3.8 | | Zhang et al. (2010) | Q | 287, 290 |
| | $4.7\times10^{-2}$ | | Zhang et al. (2010) | Q | 287, 291 |
| | $6.1\times10^{-1}$ | | Hilal et al. (2008) | Q | |
| | 3.4 | | Modarresi et al. (2007) | Q | 67 |
| | | 6500 | Kühne et al. (2005) | Q | |
| | $7.7\times10^{-1}$ | | Goss (2005) | Q | 628 |
| | $6.1\times10^{-1}$ | | Yaffe et al. (2003) | Q | 248, 249 |
| | $6.9\times10^{-1}$ | | English and Carroll (2001) | Q | 230, 231 |
| | $1.6\times10^{-2}$ | | Katritzky et al. (1998) | Q | |
| | $5.0\times10^{-1}$ | | Nirmalakhandan and Speece (1988) | Q | |
| | $5.7\times10^{-1}$ | | Duchowicz et al. (2020) | ? | 185, 21 |
| | | 5600 | Kühne et al. (2005) | ? | |
| | $5.7\times10^{-1}$ | | Abraham et al. (1990) | ? | |
| 1,1,1-trifluoro-2-propanol | $1.3\times10^{-1}$ | | Eger et al. (1999) | M | 14 |
| $CF_3CHOHCH_3$ | $4.5\times10^{-1}$ | 6300 | Rochester and Symonds (1973) | M | |
| [374-01-6] | $4.6\times10^{-1}$ | | Keshavarz et al. (2022) | Q | |
| GILIYJDBJZWGBG-UHFFFAOYSA-N | 1.1 | | Duchowicz et al. (2020) | Q | 184 |
| | $3.1\times10^{-1}$ | | Raventos-Duran et al. (2010) | Q | 242, 243 |
| | $1.6\times10^{-1}$ | | Raventos-Duran et al. (2010) | Q | 244 |
| | $2.5\times10^{-1}$ | | Raventos-Duran et al. (2010) | Q | 245 |
| | $2.2\times10^{-1}$ | | Hilal et al. (2008) | Q | |
| | $4.9\times10^{-1}$ | | Modarresi et al. (2007) | Q | 67 |
| | | 6900 | Kühne et al. (2005) | Q | |
| | $5.2\times10^{-1}$ | | Nirmalakhandan and Speece (1988) | Q | |
| | $4.5\times10^{-1}$ | | Duchowicz et al. (2020) | ? | 185, 21 |
| | | 6300 | Kühne et al. (2005) | ? | |



Table A5.1: Organic fluorine (... continued)

| Substance Formula (Trivial Name) [CAS Registry Number] InChIKey | $H_s^{cp}$ (at $T^{\ominus}$) $\left[\dfrac{\mathrm{mol}}{\mathrm{m}^3\,\mathrm{Pa}}\right]$ | $\dfrac{\mathrm{d}\ln H_s^{cp}}{\mathrm{d}(1/T)}$ [K] | Reference | Type | Note |
|---|---|---|---|---|---|
| 2,2,3,3-tetrafluoro-1-propanol | 1.4 | 7000 | Burkholder et al. (2019) | L | |
| $CHF_2CF_2CH_2OH$ | 1.4 | 7000 | Burkholder et al. (2015) | L | |
| [76-37-9] | 1.4 | 7000 | Sander et al. (2011) | L | |
| NBUKAOOFKZFCGD-UHFFFAOYSA-N | 1.4 | 7000 | Chen et al. (2003) | M | |
| | $7.5 \times 10^{-1}$ | | Eger et al. (1999) | M | 14 |
| | 1.6 | 6700 | Rochester and Symonds (1973) | M | |
| | $4.6 \times 10^{-1}$ | | Keshavarz et al. (2022) | Q | |
| | $8.3 \times 10^{-1}$ | | Duchowicz et al. (2020) | Q | |
| | 1.2 | | Raventos-Duran et al. (2010) | Q | 242, 243 |
| | $3.9 \times 10^{-1}$ | | Raventos-Duran et al. (2010) | Q | 244 |
| | $1.2 \times 10^{-1}$ | | Raventos-Duran et al. (2010) | Q | 245 |
| | $6.0 \times 10^{-1}$ | | Hilal et al. (2008) | Q | |
| | 1.0 | | Modarresi et al. (2007) | Q | 67 |
| | | 6900 | Kühne et al. (2005) | Q | |
| | $3.7 \times 10^{-1}$ | | Nirmalakhandan and Speece (1988) | Q | |
| | 1.6 | | Duchowicz et al. (2020) | ? | 185, 21 |
| | | 6600 | Kühne et al. (2005) | ? | |
| 2,2,3,3,3-pentafluoro-1-propanol | $1.4 \times 10^{-1}$ | 4300 | Burkholder et al. (2019) | L | |
| $CF_3CF_2CH_2OH$ | $1.4 \times 10^{-1}$ | 4300 | Burkholder et al. (2015) | L | |
| [422-05-9] | $1.4 \times 10^{-1}$ | 4300 | Sander et al. (2011) | L | |
| PSQZJKGXDGNDFP-UHFFFAOYSA-N | $1.4 \times 10^{-1}$ | 4300 | Chen et al. (2003) | M | |
| | $6.9 \times 10^{-2}$ | | Eger et al. (1999) | M | 14 |
| | $4.5 \times 10^{-1}$ | 6000 | Rochester and Symonds (1973) | M | |
| | $4.6 \times 10^{-1}$ | | Keshavarz et al. (2022) | Q | |
| | $1.8 \times 10^{-1}$ | | Duchowicz et al. (2020) | Q | 184 |
| | $2.0 \times 10^{-1}$ | | Raventos-Duran et al. (2010) | Q | 242, 243 |
| | $2.5 \times 10^{-1}$ | | Raventos-Duran et al. (2010) | Q | 244 |
| | $6.2 \times 10^{-2}$ | | Raventos-Duran et al. (2010) | Q | 245 |
| | $2.3 \times 10^{-1}$ | | Hilal et al. (2008) | Q | |
| | $4.8 \times 10^{-1}$ | | Modarresi et al. (2007) | Q | 67 |
| | | 6800 | Kühne et al. (2005) | Q | |
| | $4.4 \times 10^{-1}$ | | Duchowicz et al. (2020) | ? | 185, 21 |
| | | 6000 | Kühne et al. (2005) | ? | |
| 1,1,1,3,3,3-hexafluoro-2-propanol | $1.0 \times 10^{-1}$ | | Eger et al. (1999) | M | 14 |
| $CF_3CHOHCF_3$ | $2.4 \times 10^{-1}$ | 6700 | Rochester and Symonds (1973) | M | |
| [920-66-1] | $4.6 \times 10^{-1}$ | | Keshavarz et al. (2022) | Q | |
| BYEAHWXPCBROCE-UHFFFAOYSA-N | 2.2 | | Duchowicz et al. (2020) | Q | |
| | $1.6 \times 10^{-1}$ | | Raventos-Duran et al. (2010) | Q | 242, 243 |
| | $2.0 \times 10^{-2}$ | | Raventos-Duran et al. (2010) | Q | 244 |
| | $3.1 \times 10^{-2}$ | | Raventos-Duran et al. (2010) | Q | 245 |
| | $2.5 \times 10^{-2}$ | | Hilal et al. (2008) | Q | |
| | $2.6 \times 10^{-1}$ | | Modarresi et al. (2007) | Q | 67 |
| | | 6800 | Kühne et al. (2005) | Q | |
| | $2.4 \times 10^{-1}$ | | Goss (2005) | Q | 629 |
| | $2.3 \times 10^{-1}$ | | Nirmalakhandan and Speece (1988) | Q | |
| | $2.3 \times 10^{-1}$ | | Duchowicz et al. (2020) | ? | 185, 21 |




Table A5.1: Organic fluorine (...continued)

| Substance<br>Formula<br>(Trivial Name)<br>[CAS Registry Number]<br>InChIKey | $H_s^{cp}$<br>(at $T^\ominus$)<br>$\left[\dfrac{\text{mol}}{\text{m}^3\,\text{Pa}}\right]$ | $\dfrac{\text{d}\ln H_s^{cp}}{\text{d}(1/T)}$<br><br>[K] | Reference | Type | Note |
|---|---|---|---|---|---|
| | | 6700 | Kühne et al. (2005) | ? | |
| | $2.3\times10^{-1}$ | | Abraham et al. (1990) | ? | |
| trifluoroacetylfluoride | $3.0\times10^{-2}$ | | Mirabel et al. (1996) | M | |
| $CF_3COF$ | $9.5\times10^{-3}$ | | De Bruyn et al. (1995a) | M | 449 |
| [354-34-7] | $3.0\times10^{-2}$ | | George et al. (1994b) | M | 630 |
| DCEPGADSNJKOJK-UHFFFAOYSA-N | $1.2\times10^{-2}$ | | Keshavarz et al. (2022) | Q | |
| | $4.3\times10^{-3}$ | | Duchowicz et al. (2020) | Q | |
| | $3.0\times10^{-2}$ | | Duchowicz et al. (2020) | ? | 185, 21 |
| 1-fluoro-2-propanone | $4.6\times10^{-1}$ | | Burkholder et al. (2019) | L | |
| $CH_2FCOCH_3$ | $4.6\times10^{-1}$ | | Burkholder et al. (2015) | L | |
| (fluoroacetone) | $4.6\times10^{-1}$ | | O'Farrell and Waghorne (2010) | M | |
| [430-51-3] | | | | | |
| MSWVMWGCNZQPIA-UHFFFAOYSA-N | | | | | |
| 1,1,1-trifluoro-2-propanone | 1.4 | 8900 | Burkholder et al. (2019) | L | |
| $CF_3COCH_3$ | 1.4 | 8900 | Burkholder et al. (2015) | L | |
| (1,1,1-trifluoroacetone) | 1.4 | 8900 | Sander et al. (2011) | L | |
| [421-50-1] | 1.4 | 8900 | Betterton (1991) | M | |
| FHUDAMLDXFJHJE-UHFFFAOYSA-N | 1.4 | | Keshavarz et al. (2022) | Q | |
| | $1.8\times10^{-2}$ | | Duchowicz et al. (2020) | Q | |
| | 4.9 | | Raventos-Duran et al. (2010) | Q | 242, 243 |
| | $9.9\times10^{-2}$ | | Raventos-Duran et al. (2010) | Q | 244 |
| | $2.5\times10^{-2}$ | | Raventos-Duran et al. (2010) | Q | 245 |
| | $4.4\times10^{-2}$ | | Modarresi et al. (2007) | Q | 67 |
| | 1.3 | | Duchowicz et al. (2020) | ? | 185, 21 |
| 1,1-difluoro-2-methoxyethane | $1.3\times10^{-2}$ | | Duchowicz et al. (2020) | V | 186 |
| $C_3H_6F_2O$ | $2.2\times10^{-2}$ | | Duchowicz et al. (2020) | Q | |
| [461-57-4] | | | | | |
| CRGZRXUKXVTRNO-UHFFFAOYSA-N | | | | | |
| fluoroethanoic acid | $8.0\times10^2$ | | Burkholder et al. (2019) | L | |
| $CH_2FCOOH$ | $8.0\times10^2$ | | Burkholder et al. (2015) | L | |
| (fluoroacetic acid) | $8.0\times10^2$ | | Sander et al. (2011) | L | |
| [144-49-0] | $8.0\times10^2$ | | Bowden et al. (1998a) | M | |
| QEWYKACRFQMRMB-UHFFFAOYSA-N | $5.6\times10^2$ | | Keshavarz et al. (2022) | Q | |
| | $6.7\times10^1$ | | Duchowicz et al. (2020) | Q | 184 |
| | $6.2\times10^2$ | | Raventos-Duran et al. (2010) | Q | 271, 243 |
| | $9.9\times10^2$ | | Raventos-Duran et al. (2010) | Q | 244 |
| | 9.9 | | Raventos-Duran et al. (2010) | Q | 245 |
| | $5.4\times10^2$ | | Hilal et al. (2008) | Q | |
| | $8.0\times10^2$ | | Duchowicz et al. (2020) | ? | 185, 21 |




Table A5.1: Organic fluorine (...continued)

| Substance Formula (Trivial Name) [CAS Registry Number] InChIKey | $H_s^{cp}$ (at $T^\ominus$) $\left[\dfrac{\mathrm{mol}}{\mathrm{m}^3\,\mathrm{Pa}}\right]$ | $\dfrac{\mathrm{d}\ln H_s^{cp}}{\mathrm{d}(1/T)}$ [K] | Reference | Type | Note |
|---|---|---|---|---|---|
| difluoroethanoic acid | $3.0\times10^2$ | 6900 | Burkholder et al. (2019) | L | |
| $CHF_2COOH$ | $3.0\times10^2$ | 6900 | Burkholder et al. (2015) | L | |
| (difluoroacetic acid) | $3.0\times10^2$ | 6900 | Sander et al. (2011) | L | |
| [381-73-7] | $3.0\times10^2$ | 6900 | Bowden et al. (1998a) | M | |
| PBWZKZYHONABLN-UHFFFAOYSA-N | $5.6\times10^2$ | | Keshavarz et al. (2022) | Q | |
| | $8.5\times10^1$ | | Duchowicz et al. (2020) | Q | 299 |
| | $4.9\times10^2$ | | Raventos-Duran et al. (2010) | Q | 242, 243 |
| | $9.9\times10^1$ | | Raventos-Duran et al. (2010) | Q | 244 |
| | 4.9 | | Raventos-Duran et al. (2010) | Q | 245 |
| | $7.2\times10^1$ | | Hilal et al. (2008) | Q | |
| | | 7700 | Kühne et al. (2005) | Q | |
| | $3.0\times10^2$ | | Duchowicz et al. (2020) | ? | 185, 21 |
| | | 6900 | Kühne et al. (2005) | ? | |
| trifluoroethanoic acid | $5.7\times10^1$ | 4100 | Burkholder et al. (2019) | L | |
| $CF_3COOH$ | $5.7\times10^1$ | 4100 | Burkholder et al. (2015) | L | |
| (trifluoroacetic acid) | $8.9\times10^1$ | 9300 | Sander et al. (2011) | L | |
| [76-05-1] | $5.7\times10^1$ | 4100 | Kutsuna and Horia (2008) | M | |
| DTQVDTLACAAQTR-UHFFFAOYSA-N | $8.3\times10^1$ | | Kwan (2001) | M | 631 |
| | $8.8\times10^1$ | 9300 | Bowden et al. (1996) | M | |
| | $5.6\times10^2$ | | Keshavarz et al. (2022) | Q | |
| | 2.3 | | Abusallout et al. (2022) | Q | 632 |
| | $1.9\times10^1$ | | Duchowicz et al. (2020) | Q | 184 |
| | $6.2\times10^1$ | | Raventos-Duran et al. (2010) | Q | 242, 243 |
| | $1.6\times10^{-1}$ | | Raventos-Duran et al. (2010) | Q | 244 |
| | 2.5 | | Raventos-Duran et al. (2010) | Q | 245 |
| | 2.3 | | Zhang et al. (2010) | Q | 287, 288 |
| | $1.6\times10^{-1}$ | | Zhang et al. (2010) | Q | 287, 289 |
| | 8.0 | | Zhang et al. (2010) | Q | 287, 290 |
| | 3.9 | | Zhang et al. (2010) | Q | 287, 291 |
| | $4.0\times10^{-1}$ | | Hilal et al. (2008) | Q | |
| | $6.3\times10^1$ | | Modarresi et al. (2007) | Q | 67 |
| | | 7700 | Kühne et al. (2005) | Q | |
| | $8.9\times10^1$ | | Duchowicz et al. (2020) | ? | 185, 21 |
| | | 9400 | Kühne et al. (2005) | ? | |
| perfluoropropanoic acid | 8.8 | | Kwan (2001) | M | 631 |
| $C_3HF_5O_2$ | $4.3\times10^{-1}$ | | Abusallout et al. (2022) | Q | 632 |
| [422-64-0] | | | | | |
| LRMSQVBRUNSOJL-UHFFFAOYSA-N | | | | | |
| perfluorobutanoic acid | $8.1\times10^{-1}$ | | Kwan (2001) | M | 631 |
| $C_4HF_7O_2$ | $8.2\times10^{-2}$ | | Abusallout et al. (2022) | Q | 632 |
| [375-22-4] | $8.2\times10^{-2}$ | | Zhang et al. (2010) | Q | 287, 288 |
| YPJUNDFVDDCYIH-UHFFFAOYSA-N | $7.2\times10^{-1}$ | | Zhang et al. (2010) | Q | 287, 289 |
| | $2.5\times10^{-1}$ | | Zhang et al. (2010) | Q | 287, 290 |
| | $6.4\times10^{-1}$ | | Zhang et al. (2010) | Q | 287, 291 |





Table A5.1: Organic fluorine (...continued)

| Substance Formula (Trivial Name) [CAS Registry Number] InChIKey | $H_s^{cp}$ (at $T^{\ominus}$) $\left[\dfrac{\text{mol}}{\text{m}^3\,\text{Pa}}\right]$ | $\dfrac{\text{d}\ln H_s^{cp}}{\text{d}(1/T)}$ [K] | Reference | Type | Note |
|---|---|---|---|---|---|
| perfluoropentanoic acid $C_5HF_9O_2$ [2706-90-3] CXZGQIAOTKWCDB-UHFFFAOYSA-N | $6.7\times10^{-1}$ $1.6\times10^{-2}$ | | Kwan (2001) Abusallout et al. (2022) | M Q | 631 632 |
| perfluorohexanoic acid $C_6HF_{11}O_2$ [307-24-4] PXUULQAPEKKVAH-UHFFFAOYSA-N | 1.1 $3.0\times10^{-3}$ $4.4\times10^{-1}$ $1.2\times10^{-1}$ | | Kwan (2001) Abusallout et al. (2022) Arp et al. (2006) Arp et al. (2006) | M Q Q Q | 631 632 633 634 |
| perfluoroheptanoic acid $C_7HF_{13}O_2$ [375-85-9] ZWBAMYVPMDSJGQ-UHFFFAOYSA-N | 1.7 $5.8\times10^{-4}$ $5.7\times10^{-4}$ $5.0\times10^{-2}$ $2.2\times10^{-2}$ $5.6\times10^{-3}$ $1.8\times10^{-1}$ $5.7\times10^{-2}$ | | Kwan (2001) Abusallout et al. (2022) Zhang et al. (2010) Zhang et al. (2010) Zhang et al. (2010) Zhang et al. (2010) Arp et al. (2006) Arp et al. (2006) | M Q Q Q Q Q Q Q | 631 632 287, 288 287, 289 287, 290 287, 291 633 634 |
| pentadecafluorooctanoic acid $C_8HF_{15}O_2$ (perfluorooctanoic acid; PFOA) [335-67-1] SNGREZUHAYWORS-UHFFFAOYSA-N | $4.9\times10^{-2}$ $4.0\times10^{-1}$ 2.8 $1.1\times10^{-4}$ $1.1\times10^{-4}$ $1.0\times10^{-2}$ $1.2\times10^{-2}$ $1.1\times10^{-3}$ $1.1\times10^{-4}$ $1.0\times10^{-2}$ $2.1\times10^{-2}$ $1.1\times10^{-3}$ $9.5\times10^{-2}$ $2.0\times10^{-2}$ | | Kutsuna and Hori (2008) Li et al. (2007) Kwan (2001) Abusallout et al. (2022) Zhang et al. (2010) Zhang et al. (2010) Zhang et al. (2010) Zhang et al. (2010) Zhang et al. (2010) Zhang et al. (2010) Zhang et al. (2010) Zhang et al. (2010) Arp et al. (2006) Arp et al. (2006) | M M M Q Q Q Q Q Q Q Q Q Q Q | 631 632 287, 288 287, 289 287, 290 287, 291 287, 288 287, 289 287, 290 287, 291 633 634 |
| perfluorononanoic acid $C_9HF_{17}O_2$ [375-95-1] UZUFPBIDKMEQEQ-UHFFFAOYSA-N | $4.3\times10^{-2}$ $5.3\times10^{-3}$ | | Arp et al. (2006) Arp et al. (2006) | Q Q | 633 634 |
| 2H,2H-perfluorodecanoic acid $C_{10}H_3F_{17}O_2$ (8:2 FTCA) [27854-31-5] XTBXSCIWOVSSGB-UHFFFAOYSA-N | $5.8\times10^{-4}$ | | Abusallout et al. (2022) | M | |
| perfluorodecanoic acid $C_{10}HF_{19}O_2$ [335-76-2] PCIUEQPBYFRTEM-UHFFFAOYSA-N | $2.5\times10^{-2}$ $1.1\times10^{-3}$ | | Arp et al. (2006) Arp et al. (2006) | Q Q | 633 634 |



Table A5.1: Organic fluorine (. . . continued)

| Substance Formula (Trivial Name) [CAS Registry Number] InChIKey | $H_s^{cp}$ (at $T^\ominus$) $\left[\dfrac{\text{mol}}{\text{m}^3\,\text{Pa}}\right]$ | $\dfrac{\text{d}\ln H_s^{cp}}{\text{d}(1/T)}$ [K] | Reference | Type | Note |
|---|---|---|---|---|---|
| perfluoroundecanoic acid $C_{11}HF_{21}O_2$ [2058-94-8] SIDINRCMMRKXGQ-UHFFFAOYSA-N | $1.3\times10^{-2}$ $1.9\times10^{-4}$ | | Arp et al. (2006) Arp et al. (2006) | Q Q | 633 634 |
| perfluorododecanoic acid $C_{12}HF_{23}O_2$ [307-55-1] CXGONMQFMIYUJR-UHFFFAOYSA-N | $6.4\times10^{-3}$ | | Plassmann et al. (2011) | E | |
| perfluorotetradecanoic acid $C_{14}HF_{27}O_2$ [376-06-7] RUDINRUXCKIXAJ-UHFFFAOYSA-N | $1.6\times10^{-3}$ | | Plassmann et al. (2011) | E | |
| 1,1,1,3,3,3-hexafluoro-2-propanone $C_3F_6O$ [684-16-2] VBZWSGALLODQNC-UHFFFAOYSA-N | $3.2\times10^{-3}$ | | HSDB (2015) | Q | 99 |
| desflurane $C_3H_2F_6O$ [57041-67-5] DPYMFVXJLLWWEU-UHFFFAOYSA-N | $1.4\times10^{-4}$ $9.0\times10^{-5}$ | | HSDB (2015) Abraham and Weathersby (1994) | Q ? | 99 21 |
| sevoflurane $C_4H_3F_7O$ [28523-86-6] DFEYYRMXOJXZRJ-UHFFFAOYSA-N | $5.2\times10^{-5}$ $1.5\times10^{-4}$ | | HSDB (2015) Abraham and Weathersby (1994) | Q ? | 99 21 |
| ethyl 2,2,2-trifluoroethyl ether $C_4H_7F_3O$ [461-24-5] ZKNHDJMXIUOHLX-UHFFFAOYSA-N | $7.2\times10^{-4}$ | | Hilal et al. (2008) | Q | |
| iso-indoklon $C_4H_4F_6O$ (1,1,1,3,3,3-hexafluoro-2-methoxypropane) [13171-18-1] VNXYDFNVQBICRO-UHFFFAOYSA-N | $5.6\times10^{-5}$ | | Abraham and Weathersby (1994) | ? | 21 |
| di(2,2,2-trifluoroethyl) ether $C_4H_4F_6O$ (flurothyl) [333-36-8] KGPPDNUWZNWPSI-UHFFFAOYSA-N | $9.2\times10^{-3}$ $3.0\times10^{-4}$ | -390 | Fukuchi et al. (2002) Abraham and Weathersby (1994) | V ? | 33 21 |



Table A5.1: Organic fluorine (... continued)

| Substance Formula (Trivial Name) [CAS Registry Number] InChIKey | $H_s^{cp}$ (at $T^\ominus$) $\left[\dfrac{\mathrm{mol}}{\mathrm{m}^3\,\mathrm{Pa}}\right]$ | $\dfrac{\mathrm{d}\ln H_s^{cp}}{\mathrm{d}(1/T)}$ [K] | Reference | Type | Note |
|---|---|---|---|---|---|
| (2,2,2-trifluoroethoxy)-ethene | $5.4\times10^{-4}$ | 4000 | Fogg and Sangster (2003) | L | |
| $CF_3CH_2OCHCH_2$ | $3.3\times10^{-4}$ | | Steward et al. (1973) | L | 14 |
| (fluoroxene; fluroxene) | $5.5\times10^{-4}$ | 4000 | Allott et al. (1973) | L | |
| [406-90-6] | $5.5\times10^{-4}$ | 4300 | Smith et al. (1981b) | M | |
| DLEGDLSLRSOURQ-UHFFFAOYSA-N | $3.2\times10^{-4}$ | | Stoelting and Longshore (1972) | M | 14 |
| | $3.3\times10^{-4}$ | | Munson et al. (1964) | M | 14 |
| | $9.5\times10^{-5}$ | | Hilal et al. (2008) | Q | |
| | $3.6\times10^{-4}$ | | Abraham and Weathersby (1994) | ? | 21 |
| | $5.1\times10^{-4}$ | | Abraham et al. (1990) | ? | |
| 2,2,2-trifluoroethyl methanoate | $5.4\times10^{-3}$ | 4700 | Burkholder et al. (2019) | L | |
| $HCOOCH_2CF_3$ | $5.4\times10^{-3}$ | 4700 | Burkholder et al. (2015) | L | |
| [32042-38-9] | $5.4\times10^{-3}$ | 4700 | Sander et al. (2011) | L | |
| CAFROQYMUICGNO-UHFFFAOYSA-N | $5.4\times10^{-3}$ | 4700 | Kutsuna et al. (2005) | M | |
| 2,2,2-trifluoroethyl ethanoate | $5.5\times10^{-3}$ | 5200 | Burkholder et al. (2019) | L | 635 |
| $CH_3COOCH_2CF_3$ | $5.5\times10^{-3}$ | 5200 | Burkholder et al. (2015) | L | 636 |
| [406-95-1] | $5.5\times10^{-3}$ | 5200 | Sander et al. (2011) | L | |
| ZOWSJJBOQDKOHI-UHFFFAOYSA-N | $5.7\times10^{-3}$ | 5300 | Kutsuna et al. (2004) | M | |
| | | 6400 | Kühne et al. (2005) | Q | |
| | | 5500 | Kühne et al. (2005) | ? | |
| trifluoroethanoic acid, methyl ester | $1.1\times10^{-3}$ | 5300 | Burkholder et al. (2019) | L | 637, 638 |
| $CF_3COOCH_3$ | $1.1\times10^{-3}$ | 5300 | Burkholder et al. (2015) | L | 639, 640 |
| (methyl trifluoroacetate) | $1.1\times10^{-3}$ | 5300 | Sander et al. (2011) | L | 641 |
| [431-47-0] | $1.2\times10^{-3}$ | 4900 | Kutsuna et al. (2004) | M | |
| VMVNZNXAVJHNDJ-UHFFFAOYSA-N | | 6100 | Kühne et al. (2005) | Q | |
| | | 5800 | Kühne et al. (2005) | ? | |
| trifluoroethanoic acid, ethyl ester | $8.9\times10^{-4}$ | 4900 | Burkholder et al. (2019) | L | |
| $CF_3COOC_2H_5$ | $8.9\times10^{-4}$ | 4900 | Burkholder et al. (2015) | L | |
| (ethyl trifluoroacetate) | $8.9\times10^{-4}$ | 4900 | Sander et al. (2011) | L | |
| [383-63-1] | $7.1\times10^{-4}$ | 4900 | Kutsuna and Kaneyasu (2021) | M | |
| STSCVKRWJPWALQ-UHFFFAOYSA-N | $8.9\times10^{-4}$ | 4900 | Kutsuna et al. (2005) | M | |
| trifluoro(trifluoromethyl)-oxirane | $9.3\times10^{-6}$ | 2400 | Clever et al. (2005) | C | 642, 643 |
| $C_3F_6O$ | | | | | |
| [428-59-1] | | | | | |
| PGFXOWRDDHCDTE-UHFFFAOYSA-N | | | | | |
| 3,3,4,4-pentafluorobutan-1-ol | $5.1\times10^{-2}$ | | Zhang et al. (2010) | Q | 287, 288 |
| $C_4H_5OF_5$ | $3.7\times10^{-1}$ | | Zhang et al. (2010) | Q | 287, 289 |
| [54949-74-5] | $4.0\times10^{-2}$ | | Zhang et al. (2010) | Q | 287, 290 |
| JPMHUDBOKDBBLG-UHFFFAOYSA-N | $1.5\times10^{-3}$ | | Zhang et al. (2010) | Q | 287, 291 |
| 1,1,1,3,3,3-hexafluoro-2-methyl-2-propanol $C_4H_4F_6O$ [1515-14-6] FQDXJYBXPOMIBX-UHFFFAOYSA-N | $1.8\times10^{-2}$ | | Eger et al. (1999) | M | 14 |





Table A5.1: Organic fluorine (...continued)

| Substance Formula (Trivial Name) [CAS Registry Number] InChIKey | $H_s^{cp}$ (at $T^\ominus$) $\left[\dfrac{\text{mol}}{\text{m}^3\,\text{Pa}}\right]$ | $\dfrac{\text{d}\ln H_s^{cp}}{\text{d}(1/T)}$ [K] | Reference | Type | Note |
|---|---|---|---|---|---|
| 2,2,3,4,4,4-hexafluoro-1-butanol $C_4H_4F_6O$ [382-31-0] LVFXLZRISXUAIL-UHFFFAOYSA-N | $3.2\times10^{-1}$ | | Eger et al. (1999) | M | 14 |
| 2,2,3,3,4,4,4-heptafluoro-1-butanol $C_4H_3F_7O$ [375-01-9] WXJFKAZDSQLPBX-UHFFFAOYSA-N | $2.1\times10^{-2}$ | | Eger et al. (1999) | M | 14 |
| 3,3,4,4,5,5,5-heptafluoro-2-pentanol $C_5H_5F_7O$ [375-14-4] RBPHBIMHZSTIDT-UHFFFAOYSA-N | $9.0\times10^{-3}$ | | Eger et al. (1999) | M | 14 |
| 2,2,3,3,4,4,5,5-octafluoro-1-pentanol $C_5H_4F_8O$ [355-80-6] JUGSKHLZINSXPQ-UHFFFAOYSA-N | $2.5\times10^{-1}$ | | Eger et al. (1999) | M | 14 |
| 1,1,1,2,2,3,3,4,4-nonafluoro-4-methoxybutane $C_5H_3F_9O$ [163702-07-6] OKIYQFLILPKULA-UHFFFAOYSA-N | $9.9\times10^{-6}$ $1.3\times10^{-5}$ $8.4\times10^{-6}$ $3.9\times10^{-6}$ | | Zhang et al. (2010) Zhang et al. (2010) Zhang et al. (2010) Zhang et al. (2010) | Q Q Q Q | 287, 288 287, 289 287, 290 287, 291 |
| 1-ethoxy-1,1,2,3,3,3-hexafluoro-2-(trifluoromethyl)propane $C_6H_5F_9O$ [163702-06-5] SQEGLLMNIBLLNQ-UHFFFAOYSA-N | $7.5\times10^{-6}$ $4.7\times10^{-5}$ $8.0\times10^{-6}$ $3.3\times10^{-6}$ | | Zhang et al. (2010) Zhang et al. (2010) Zhang et al. (2010) Zhang et al. (2010) | Q Q Q Q | 287, 288 287, 289 287, 290 287, 291 |
| 1H,1H,2H,2H-perfluorohexan-1-ol $C_6H_5F_9O$ (4:2 FTOH) [2043-47-2] JCMNMOBHVPONLD-UHFFFAOYSA-N | $1.3\times10^{-3}$ $6.6\times10^{-3}$ $1.3\times10^{-2}$ $6.1\times10^{-5}$ $5.6\times10^{-1}$ $1.8\times10^{-3}$ $1.8\times10^{-3}$ $1.3\times10^{-1}$ $8.2\times10^{-3}$ $2.4\times10^{-4}$ $4.3\times10^{-4}$ $3.1\times10^{-5}$ $7.2\times10^{-3}$ | 4500 5400 7200 | Abusallout et al. (2022) Wu and Chang (2011) Goss et al. (2006) Lei et al. (2004) Wu and Chang (2011) Abusallout et al. (2022) Zhang et al. (2010) Zhang et al. (2010) Zhang et al. (2010) Zhang et al. (2010) Arp et al. (2006) Arp et al. (2006) Goss et al. (2006) | M M M M V Q Q Q Q Q Q Q Q | 11 327 632 287, 288 287, 289 287, 290 287, 291 633 634 |



Table A5.1: Organic fluorine (...continued)

| Substance Formula (Trivial Name) [CAS Registry Number] InChIKey | $H_s^{cp}$ (at $T^\ominus$) $\left[\dfrac{\text{mol}}{\text{m}^3\,\text{Pa}}\right]$ | $\dfrac{\text{d}\ln H_s^{cp}}{\text{d}(1/T)}$ [K] | Reference | Type | Note |
|---|---|---|---|---|---|
| 2,2,3,3,4,4,5,5,6,6,6-undecafluoro-1-hexanol<br>$C_6H_3F_{11}O$<br>[423-46-1]<br>QZFZPVVDBGXQTB-UHFFFAOYSA-N | $1.7\times10^{-3}$ | | Eger et al. (1999) | M | 14 |
| 1-ethoxy-1,1,2,2,3,3,4,4,4-nonafluorobutane<br>$C_6H_5F_9O$<br>[163702-05-4]<br>DFUYAWQUODQGFF-UHFFFAOYSA-N | $7.5\times10^{-6}$<br>$1.2\times10^{-5}$<br>$7.5\times10^{-6}$<br>$3.0\times10^{-6}$ | | Zhang et al. (2010)<br>Zhang et al. (2010)<br>Zhang et al. (2010)<br>Zhang et al. (2010) | Q<br>Q<br>Q<br>Q | 287, 288<br>287, 289<br>287, 290<br>287, 291 |
| 2,2,3,3,4,4,5,5,6,6,7,7-dodecafluoro-1-heptanol<br>$C_7H_4F_{12}O$<br>[335-99-9]<br>BYKNGMLDSIEFFG-UHFFFAOYSA-N | $6.4\times10^{-2}$ | | Eger et al. (1999) | M | 14 |
| 2,2,3,3,4,4,5,5,6,6,7,7,7-tridecafluoro-1-heptanol<br>$C_7H_3F_{13}O$<br>[375-82-6]<br>STLNAVFVCIRZLL-UHFFFAOYSA-N | $6.0\times10^{-4}$ | | Eger et al. (1999) | M | 14 |
| 1H,1H,2H,2H-perfluoro-1-octanol<br>$C_8H_5F_{13}O$<br>(6:2 FTOH)<br>[647-42-7]<br>GRJRKPMIRMSBNK-UHFFFAOYSA-N | $3.3\times10^{-4}$<br>$1.7\times10^{-4}$<br>$1.5\times10^{-3}$<br>$8.5\times10^{-5}$<br>$9.4\times10^{-4}$<br>$3.9\times10^{-1}$<br>$6.6\times10^{-5}$<br>$6.5\times10^{-5}$<br>$9.5\times10^{-3}$<br>$3.4\times10^{-3}$<br>$9.9\times10^{-6}$<br>$2.8\times10^{-4}$<br>$1.8\times10^{-5}$<br>$1.8\times10^{-3}$ | 4700<br>2600<br><br>7000<br><br><br><br><br><br><br><br><br><br>8000 | Abusallout et al. (2022)<br>Wu and Chang (2011)<br>Goss et al. (2006)<br>Lei et al. (2004)<br>Eger et al. (1999)<br>Wu and Chang (2011)<br>Abusallout et al. (2022)<br>Zhang et al. (2010)<br>Zhang et al. (2010)<br>Zhang et al. (2010)<br>Zhang et al. (2010)<br>Arp et al. (2006)<br>Arp et al. (2006)<br>Goss et al. (2006) | M<br>M<br>M<br>M<br>M<br>V<br>Q<br>Q<br>Q<br>Q<br>Q<br>Q<br>Q<br>Q | <br>11<br><br>327<br>14<br><br>632<br>287, 288<br>287, 289<br>287, 290<br>287, 291<br>633<br>634<br> |
| 2,2,3,3,4,4,5,5,6,6,7,7,8,8,8-pentadecafluoro-1-octanol<br>$C_8H_3F_{15}O$<br>[307-30-2]<br>PJDOLCGOTSNFJM-UHFFFAOYSA-N | $2.3\times10^{-4}$ | | Eger et al. (1999) | M | 14 |
| 3-ethoxyperfluoro(2-methylhexane)<br>$C_9H_5F_{15}O$<br>[297730-93-9]<br>HHBBIOLEJRWIGU-UHFFFAOYSA-N | $1.9\times10^{-8}$ | | Ebert et al. (2023) | ? | 365 |





Table A5.1: Organic fluorine (...continued)

| Substance Formula (Trivial Name) [CAS Registry Number] InChIKey | $H_s^{cp}$ (at $T^{\ominus}$) $\left[\dfrac{\text{mol}}{\text{m}^3\,\text{Pa}}\right]$ | $\dfrac{\text{d}\ln H_s^{cp}}{\text{d}(1/T)}$ [K] | Reference | Type | Note |
|---|---|---|---|---|---|
| methyl perfluoro(8-(fluoroformyl)-5-methyl-4,7-dioxanonanoate) | $5.8\times10^{-2}$ | | Zhang et al. (2010) | Q | 287, 288 |
| $C_{10}H_3F_{15}O_5$ | $5.1\times10^{-4}$ | | Zhang et al. (2010) | Q | 287, 289 |
| [69116-73-0] | $2.6\times10^{-4}$ | | Zhang et al. (2010) | Q | 287, 290 |
| JOMJXRTUQWIHQD-UHFFFAOYSA-N | $1.3\times10^{-1}$ | | Zhang et al. (2010) | Q | 287, 291 |
| 3,3,4,4,5,5,6,6,6-nonafluorohexyl methacrylate | $3.4\times10^{-5}$ | | Zhang et al. (2010) | Q | 287, 288 |
| $C_{10}H_9F_9O_2$ | $1.6\times10^{-3}$ | | Zhang et al. (2010) | Q | 287, 289 |
| [1799-84-4] | $6.5\times10^{-4}$ | | Zhang et al. (2010) | Q | 287, 290 |
| TYNRPOFACABVSI-UHFFFAOYSA-N | $3.4\times10^{-5}$ | | Zhang et al. (2010) | Q | 287, 291 |
| 1H,1H,2H,2H-perfluorodecan-1-ol | $2.0\times10^{-4}$ | | Abusallout et al. (2022) | M | |
| $C_{10}H_5F_{17}O$ | $2.0\times10^{-4}$ | 3100 | Wu and Chang (2011) | M | 11 |
| (8:2 FTOH) | $1.7\times10^{-4}$ | 8800 | Lei et al. (2004) | M | 327 |
| [678-39-7] | $2.4\times10^{-1}$ | | Wu and Chang (2011) | V | |
| JJUBFBTUBACDHW-UHFFFAOYSA-N | $1.1\times10^{-4}$ | | Goss et al. (2006) | V | |
| | $2.4\times10^{-6}$ | | Abusallout et al. (2022) | Q | 632 |
| | $2.4\times10^{-6}$ | | Zhang et al. (2010) | Q | 287, 288 |
| | $2.6\times10^{-4}$ | | Zhang et al. (2010) | Q | 287, 289 |
| | $7.3\times10^{-4}$ | | Zhang et al. (2010) | Q | 287, 290 |
| | $4.3\times10^{-7}$ | | Zhang et al. (2010) | Q | 287, 291 |
| | $5.7\times10^{-5}$ | | Arp et al. (2006) | Q | 633 |
| | $1.6\times10^{-5}$ | | Arp et al. (2006) | Q | 634 |
| | $3.8\times10^{-4}$ | 8600 | Goss et al. (2006) | Q | |
| 2-methoxyperfluoro(2,5-di(propan-2-yl)oxolane) | $1.0\times10^{-8}$ | | Ebert et al. (2023) | ? | 365 |
| $C_{11}H_3F_{19}O_2$ | | | | | |
| [957209-18-6] | | | | | |
| YRGYOFYTTFLPQM-UHFFFAOYSA-N | | | | | |
| 3,3,4,4,5,5,6,6,7,7,8,8,8-tridecafluorooctyl acrylate | $1.9\times10^{-6}$ | | Zhang et al. (2010) | Q | 287, 288 |
| $C_{11}H_7F_{13}O_2$ | $1.9\times10^{-4}$ | | Zhang et al. (2010) | Q | 287, 289 |
| [17527-29-6] | $2.9\times10^{-4}$ | | Zhang et al. (2010) | Q | 287, 290 |
| VPKQPPJQTZJZDB-UHFFFAOYSA-N | $2.4\times10^{-6}$ | | Zhang et al. (2010) | Q | 287, 291 |
| 2-(perfluorohexyl)ethyl methacrylate | $1.2\times10^{-6}$ | | Zhang et al. (2010) | Q | 287, 288 |
| $C_{12}H_9F_{13}O_2$ | $1.8\times10^{-4}$ | | Zhang et al. (2010) | Q | 287, 289 |
| [2144-53-8] | $1.3\times10^{-4}$ | | Zhang et al. (2010) | Q | 287, 290 |
| CDXFIRXEAJABAZ-UHFFFAOYSA-N | $1.5\times10^{-6}$ | | Zhang et al. (2010) | Q | 287, 291 |



Table A5.1: Organic fluorine (...continued)

| Substance Formula (Trivial Name) [CAS Registry Number] InChIKey | $H_s^{cp}$ (at $T^\ominus$) $\left[\dfrac{\text{mol}}{\text{m}^3\,\text{Pa}}\right]$ | $\dfrac{\text{d}\ln H_s^{cp}}{\text{d}(1/T)}$ [K] | Reference | Type | Note |
|---|---|---|---|---|---|
| 1,1,2,2-tetrahydroperfluoro dodecanol | $1.4\times10^{-4}$ | | Abusallout et al. (2022) | M | |
| $C_{12}H_5F_{21}O$ | $1.3\times10^{-4}$ | 2700 | Wu and Chang (2011) | M | 11 |
| (10:2 FTOH) | $2.5\times10^{-1}$ | | Wu and Chang (2011) | V | |
| [865-86-1] | $9.0\times10^{-8}$ | | Abusallout et al. (2022) | Q | 632 |
| FLXYIZWPNQYPIT-UHFFFAOYSA-N | $8.6\times10^{-8}$ | | Zhang et al. (2010) | Q | 287, 288 |
| | $2.7\times10^{-6}$ | | Zhang et al. (2010) | Q | 287, 289 |
| | $1.5\times10^{-4}$ | | Zhang et al. (2010) | Q | 287, 290 |
| | $1.6\times10^{-8}$ | | Zhang et al. (2010) | Q | 287, 291 |
| | $4.6\times10^{-5}$ | | Arp et al. (2006) | Q | 633 |
| | $5.2\times10^{-5}$ | | Arp et al. (2006) | Q | 634 |
| | $1.0\times10^{-4}$ | 9600 | Goss et al. (2006) | Q | |
| | $1.0\times10^{-5}$ | | Arp et al. (2006) | E | 644 |
| 3,3,4,4,5,5,6,6,7,7,8,8,9,9,10,10,10-heptadecafluorodecyl acrylate | $1.3\times10^{-3}$ | | Abusallout et al. (2022) | M | |
| $C_{13}H_7F_{17}O_2$ | $7.0\times10^{-8}$ | | Zhang et al. (2010) | Q | 287, 288 |
| [27905-45-9] | $1.1\times10^{-5}$ | | Zhang et al. (2010) | Q | 287, 289 |
| QUKRIOLKOHUUBM-UHFFFAOYSA-N | $1.1\times10^{-4}$ | | Zhang et al. (2010) | Q | 287, 290 |
| | $9.9\times10^{-8}$ | | Zhang et al. (2010) | Q | 287, 291 |
| 3,3,4,4,5,5,6,6,7,7,8,8,9,9,10,10,10-heptadecafluorodecyl methacrylate | $4.4\times10^{-8}$ | | Zhang et al. (2010) | Q | 287, 288 |
| $C_{14}H_9F_{17}O_2$ | $1.0\times10^{-5}$ | | Zhang et al. (2010) | Q | 287, 289 |
| [1996-88-9] | $5.4\times10^{-5}$ | | Zhang et al. (2010) | Q | 287, 290 |
| HBZFBSFGXQBQTB-UHFFFAOYSA-N | $6.4\times10^{-8}$ | | Zhang et al. (2010) | Q | 287, 291 |
| 3,3,4,4,5,5,6,6,7,7,8,8,9,9,10,10,11,11,12,12,13,13,14,14,14-pentacosafluorotetradecan-1-ol | $3.1\times10^{-9}$ | | Zhang et al. (2010) | Q | 287, 288 |
| $C_{14}H_5F_{25}O$ | $1.1\times10^{-8}$ | | Zhang et al. (2010) | Q | 287, 289 |
| [39239-77-5] | $3.1\times10^{-5}$ | | Zhang et al. (2010) | Q | 287, 290 |
| QBBJBWKVSJWYQK-UHFFFAOYSA-N | $6.9\times10^{-10}$ | | Zhang et al. (2010) | Q | 287, 291 |
| 2-(perfluorodecyl)ethyl acrylate | $2.5\times10^{-9}$ | | Zhang et al. (2010) | Q | 287, 288 |
| $C_{15}H_7F_{21}O_2$ | $3.1\times10^{-7}$ | | Zhang et al. (2010) | Q | 287, 289 |
| [17741-60-5] | $2.4\times10^{-5}$ | | Zhang et al. (2010) | Q | 287, 290 |
| FIAHOPQKBBASOY-UHFFFAOYSA-N | $3.7\times10^{-9}$ | | Zhang et al. (2010) | Q | 287, 291 |
| 1,1,2,2-tetrahydroperfluoro-1-hexadecanol | $1.1\times10^{-10}$ | | Zhang et al. (2010) | Q | 287, 288 |
| $C_{16}H_5OF_{29}$ | $1.4\times10^{-11}$ | | Zhang et al. (2010) | Q | 287, 289 |
| [60699-51-6] | $6.1\times10^{-6}$ | | Zhang et al. (2010) | Q | 287, 290 |
| ZDUOTHMDVYXZBS-UHFFFAOYSA-N | $2.9\times10^{-11}$ | | Zhang et al. (2010) | Q | 287, 291 |





Table A5.1: Organic fluorine (. . . continued)

| Substance Formula (Trivial Name) [CAS Registry Number] InChIKey | $H_s^{cp}$ (at $T^{\ominus}$) $\left[\dfrac{\text{mol}}{\text{m}^3\,\text{Pa}}\right]$ | $\dfrac{\text{d}\ln H_s^{cp}}{\text{d}(1/T)}$ [K] | Reference | Type | Note |
|---|---|---|---|---|---|
| 2-(perfluorodecyl)ethyl methacrylate | $1.6\times10^{-9}$ | | Zhang et al. (2010) | Q | 287, 288 |
| $C_{16}H_9F_{21}O_2$ | $3.1\times10^{-7}$ | | Zhang et al. (2010) | Q | 287, 289 |
| [2144-54-9] | $1.1\times10^{-5}$ | | Zhang et al. (2010) | Q | 287, 290 |
| FQHLOOOXLDQLPF-UHFFFAOYSA-N | $2.4\times10^{-9}$ | | Zhang et al. (2010) | Q | 287, 291 |
| 3,3,4,4,5,5,6,6,7,7,8,8,9,9, 10,10,11,11,12,12,13,13,14,14,14- pentacosafluorotetradecyl prop-2-enoate | $9.0\times10^{-11}$ | | Zhang et al. (2010) | Q | 287, 288 |
| $C_{17}H_7F_{25}O_2$ | $5.0\times10^{-9}$ | | Zhang et al. (2010) | Q | 287, 289 |
| [34395-24-9] | $2.7\times10^{-3}$ | | Zhang et al. (2010) | Q | 287, 290 |
| SWTZSHBOMGAQKX-UHFFFAOYSA-N | $1.6\times10^{-10}$ | | Zhang et al. (2010) | Q | 287, 291 |
| profluthrin $C_{17}H_{18}F_4O_2$ [223419-20-3] AGMMRUPNXPWLGF-AATRIKPKSA-N | $2.5\times10^{-2}$ | | Ebert et al. (2023) | ? | 318 |
| 2-perfluorododecylethyl methacrylate | $5.8\times10^{-11}$ | | Zhang et al. (2010) | Q | 287, 288 |
| $C_{18}H_9F_{25}O_2$ | $5.0\times10^{-9}$ | | Zhang et al. (2010) | Q | 287, 289 |
| [6014-75-1] | $2.3\times10^{-6}$ | | Zhang et al. (2010) | Q | 287, 290 |
| LFEGLDRNIDJMKB-UHFFFAOYSA-N | $9.9\times10^{-11}$ | | Zhang et al. (2010) | Q | 287, 291 |
| 1,1,2,2-tetrahydroperfluoro-1- octadecanol | $4.1\times10^{-12}$ | | Zhang et al. (2010) | Q | 287, 288 |
| $C_{18}H_5OF_{33}$ | $6.7\times10^{-15}$ | | Zhang et al. (2010) | Q | 287, 289 |
| [65104-67-8] | $1.2\times10^{-6}$ | | Zhang et al. (2010) | Q | 287, 290 |
| UYSGWTCETIRUHO-UHFFFAOYSA-N | $1.1\times10^{-12}$ | | Zhang et al. (2010) | Q | 287, 291 |
| metofluthrin $C_{18}H_{20}F_4O_3$ [240494-70-6] KVIZNNVXXNFLMU-AATRIKPKSA-N | $1.0$ | | HSDB (2015) | V | |
| 1,1,2,2- tetrahydroperfluorohexadecyl acrylate | $3.3\times10^{-12}$ | | Zhang et al. (2010) | Q | 287, 288 |
| $C_{19}H_7F29O_2$ | $4.1\times10^{-11}$ | | Zhang et al. (2010) | Q | 287, 289 |
| [34362-49-7] | $6.5\times10^{-4}$ | | Zhang et al. (2010) | Q | 287, 290 |
| KLOHTAIHCCMZIL-UHFFFAOYSA-N | $6.9\times10^{-12}$ | | Zhang et al. (2010) | Q | 287, 291 |
| fluoxymesterone $C_{20}H_{29}FO_3$ [76-43-7] YLRFCQOZQXIBAB-YXVJBPKESA-N | $1.6\times10^{4}$ | | HSDB (2015) | Q | 99 |



Table A5.1: Organic fluorine (...continued)

| Substance<br>Formula<br>(Trivial Name)<br>[CAS Registry Number]<br>InChIKey | $H_s^{cp}$<br>(at $T^\ominus$)<br>$\left[\dfrac{\text{mol}}{\text{m}^3\,\text{Pa}}\right]$ | $\dfrac{\text{d}\ln H_s^{cp}}{\text{d}(1/T)}$<br><br>[K] | Reference | Type | Note |
|---|---|---|---|---|---|
| dexamethasone<br>$C_{22}H_{29}FO_5$<br>[50-02-2]<br>UREBDLICKHMUKA-GCMAGEFQSA-N | $1.4\times10^2$ | | HSDB (2015) | Q | 99 |
| 1,1,2,2-tetrahydroperfluoroeicosyl alcohol<br>$C_{20}H_5OF_{37}$<br>[65104-65-6]<br>FDCQNVKWWMNRQN-UHFFFAOYSA-N | $1.5\times10^{-13}$<br>$2.2\times10^{-18}$<br>$2.4\times10^{-7}$<br>$4.6\times10^{-14}$ | | Zhang et al. (2010)<br>Zhang et al. (2010)<br>Zhang et al. (2010)<br>Zhang et al. (2010) | Q<br>Q<br>Q<br>Q | 287, 288<br>287, 289<br>287, 290<br>287, 291 |
| 2-hydroxyfluorobenzene<br>$C_6H_5FO$<br>(o-fluorophenol)<br>[367-12-4]<br>HFHFGHLXUCOHLN-UHFFFAOYSA-N | 3.1<br>4.3<br>$5.1\times10^1$<br>2.3<br>2.9<br>3.1<br>$2.1\times10^2$<br>3.1 | | Abraham et al. (1994a)<br>Keshavarz et al. (2022)<br>Duchowicz et al. (2020)<br>Hilal et al. (2008)<br>Modarresi et al. (2007)<br>Yaffe et al. (2003)<br>Nirmalakhandan et al. (1997)<br>Duchowicz et al. (2020) | R<br>Q<br>Q<br>Q<br>Q<br>Q<br>Q<br>? | <br><br>299<br><br>67<br>248, 249<br><br>185, 21 |
| 4-hydroxyfluorobenzene<br>$C_6H_5FO$<br>(p-fluorophenol)<br>[371-41-5]<br>RHMPLDJJXGPMEX-UHFFFAOYSA-N | $1.4\times10^1$<br>4.3<br>$1.3\times10^2$<br>7.9<br>3.3<br>$1.4\times10^1$<br>$2.1\times10^1$<br>$2.1\times10^2$<br>$1.4\times10^1$ | | Abraham et al. (1994a)<br>Keshavarz et al. (2022)<br>Duchowicz et al. (2020)<br>Hilal et al. (2008)<br>Modarresi et al. (2007)<br>Yaffe et al. (2003)<br>English and Carroll (2001)<br>Nirmalakhandan et al. (1997)<br>Duchowicz et al. (2020) | R<br>Q<br>Q<br>Q<br>Q<br>Q<br>Q<br>Q<br>? | <br><br><br><br>67<br>248, 249<br>230, 231<br><br>185, 21 |
| 3-fluorophenol<br>$C_6H_5FO$<br>[372-20-3]<br>SJTBRFHBXDZMPS-UHFFFAOYSA-N | 9.0 | | Hilal et al. (2008) | Q | |
| 2,6-difluorophenol<br>$C_6H_4F_2O$<br>[28177-48-2]<br>CKKOVFGIBXCEIJ-UHFFFAOYSA-N | $7.0\times10^{-1}$ | | Hilal et al. (2008) | Q | |
| 4,4'-(hexafluoroisopropylidene)diphenol<br>$C_{15}H_{10}F_6O_2$<br>[1478-61-1]<br>ZFVMWEVVKGLCIJ-UHFFFAOYSA-N | $1.7\times10^4$<br>$1.7\times10^4$<br>$1.4\times10^6$<br>$2.1\times10^5$<br>$5.3\times10^3$ | | HSDB (2015)<br>Zhang et al. (2010)<br>Zhang et al. (2010)<br>Zhang et al. (2010)<br>Zhang et al. (2010) | Q<br>Q<br>Q<br>Q<br>Q | 447<br>287, 288<br>287, 289<br>287, 290<br>287, 291 |





Table A5.1: Organic fluorine (...continued)

| Substance Formula (Trivial Name) [CAS Registry Number] InChIKey | $H_s^{cp}$ (at $T^\ominus$) $\left[\dfrac{\text{mol}}{\text{m}^3\,\text{Pa}}\right]$ | $\dfrac{\text{d}\ln H_s^{cp}}{\text{d}(1/T)}$ [K] | Reference | Type | Note |
|---|---|---|---|---|---|
| flocoumafen $C_{33}H_{25}F_3O_4$ [90035-08-8] KKBGNYHHEIAGOH-UHFFFAOYSA-N | $1.4\times10^7$ | | HSDB (2015) | Q | 99 |
| 2,3,3,3-tetrafluoro-2-(trifluoromethyl)propanenitrile $C_4F_7N$ [42532-60-5] AASDJASZOZGYMM-UHFFFAOYSA-N | $1.6\times10^{-8}$ | | Ebert et al. (2023) | ? | 365 |
| 2-fluoroaniline $C_6H_6FN$ [348-54-9] FTZQXOJYPFINKJ-UHFFFAOYSA-N | 1.4 | | Ebert et al. (2023) | ? | 318 |
| 4-fluoroaniline $C_6H_6FN$ [371-40-4] KRZCOLNOCZKSDF-UHFFFAOYSA-N | 1.6 | | HSDB (2015) | Q | 447 |
| 3-(trifluoromethyl)aniline $C_7H_6F_3N$ [98-16-8] VIUDTWATMPPKEL-UHFFFAOYSA-N | $3.9\times10^{-1}$ | | Ebert et al. (2023) | ? | 316 |
| perfluorotributylamine $C_{12}F_{27}N$ [311-89-7] RVZRBWKZFJCCIB-UHFFFAOYSA-N | $1.8\times10^{-10}$ $1.8\times10^{-10}$ $3.4\times10^{-10}$ $1.8\times10^{-9}$ $2.7\times10^{-10}$ | | HSDB (2015) Zhang et al. (2010) Zhang et al. (2010) Zhang et al. (2010) Zhang et al. (2010) | Q Q Q Q Q | 99 287, 288 287, 289 287, 290 287, 291 |
| N-ethyl-1-[3-(trifluoromethyl)phenyl]-2-propanamine $C_{12}H_{16}F_3N$ (fenfluramine) [458-24-2] DBGIVFWFUFKIQN-UHFFFAOYSA-N | $3.7\times10^{-1}$ | | HSDB (2015) | Q | 99 |
| tris(undecafluoropentyl)amine $C_{15}F_{33}N$ [338-84-1] AQZYBQIAUSKCCS-UHFFFAOYSA-N | $1.2\times10^{-12}$ $1.0\times10^{-12}$ $3.4\times10^{-10}$ $2.1\times10^{-12}$ | | Zhang et al. (2010) Zhang et al. (2010) Zhang et al. (2010) Zhang et al. (2010) | Q Q Q Q | 287, 288 287, 289 287, 290 287, 291 |
| indaziflam A $C_{16}H_{20}FN_5$ [730979-19-8] YFONKFDEZLYQDH-OPQQBVKSSA-N | $1.8\times10^5$ | | Ebert et al. (2023) | ? | 318 |



Table A5.1: Organic fluorine (...continued)

| Substance Formula (Trivial Name) [CAS Registry Number] InChIKey | $H_s^{cp}$ (at $T^{\ominus}$) $\left[\dfrac{\text{mol}}{\text{m}^3\,\text{Pa}}\right]$ | $\dfrac{\text{d}\ln H_s^{cp}}{\text{d}(1/T)}$ [K] | Reference | Type | Note |
|---|---|---|---|---|---|
| indaziflam B $C_{16}H_{20}FN_5$ [730979-32-5] YFONKFDEZLYQDH-OUJBWJOFSA-N | $5.3\times10^5$ | | Ebert et al. (2023) | ? | 318 |
| cinacalcet $C_{22}H_{22}F_3N$ [226256-56-0] VDHAWDNDOKGFTD-MRXNPFEDSA-N | $4.5\times10^1$ | | HSDB (2015) | Q | 99 |
| hydramethylnon $C_{25}H_{24}F_6N_4$ [67485-29-4] IQVNEKKDSLOHHK-FNCQTZNRSA-N | 4.5 4.5 $2.7\times10^6$ | | Duchowicz et al. (2020) HSDB (2015) Duchowicz et al. (2020) | V V Q | 186 |
| 2-fluoroacetamide $C_2H_4FNO$ [640-19-7] FVTWJXMFYOXOKK-UHFFFAOYSA-N | $4.4\times10^2$ | | HSDB (2015) | Q | 99 |
| 5-fluorouracil $C_4H_3FN_2O_2$ [51-21-8] GHASVSINZRGABV-UHFFFAOYSA-N | $5.8\times10^4$ | | HSDB (2015) | Q | 99 |
| perfluoro-N-methylmorpholine $C_5F_{11}NO$ [382-28-5] PQMAKJUXOOVROI-UHFFFAOYSA-N | $6.4\times10^{-8}$ | | Ebert et al. (2023) | ? | 365 |
| 1-fluoro-2,4-dinitrobenzene $C_6H_3FN_2O_4$ [70-34-8] LOTKRQAVGJMPNV-UHFFFAOYSA-N | $1.0\times10^2$ | | HSDB (2015) | Q | 447 |
| 5-fluoro-2-nitrophenol $C_6H_4FNO_3$ [446-36-6] QQURWFRNETXFTN-UHFFFAOYSA-N | $5.0\times10^{-1}$ 5.8 | 4100 6200 | Tremp et al. (1993) Schwarzenbach et al. (1988) Kühne et al. (2005) Kühne et al. (2005) | M V Q ? | 12 12 |
| 4-nitro-3-(trifluoromethyl)phenol $C_7H_4F_3NO_3$ [88-30-2] ZEFMBAFMCSYJOO-UHFFFAOYSA-N | $5.2\times10^2$ $5.2\times10^2$ $6.7\times10^3$ $3.9\times10^4$ $1.2\times10^3$ | | HSDB (2015) Zhang et al. (2010) Zhang et al. (2010) Zhang et al. (2010) Zhang et al. (2010) | Q Q Q Q Q | 99 287, 288 287, 289 287, 290 287, 291 |
| 1-nitro-3-(trifluoromethyl)benzene $C_7H_4F_3NO_2$ [98-46-4] WHNAMGUAXHGCHH-UHFFFAOYSA-N | $5.3\times10^{-2}$ $2.0\times10^{-1}$ $5.7\times10^{-2}$ $8.2\times10^{-3}$ | | Zhang et al. (2010) Zhang et al. (2010) Zhang et al. (2010) Zhang et al. (2010) | Q Q Q Q | 287, 288 287, 289 287, 290 287, 291 |



Table A5.1: Organic fluorine (...continued)

| Substance Formula (Trivial Name) [CAS Registry Number] InChIKey | $H_s^{cp}$ (at $T^{\ominus}$) $\left[\dfrac{\mathrm{mol}}{\mathrm{m^3\,Pa}}\right]$ | $\dfrac{\mathrm{d}\ln H_s^{cp}}{\mathrm{d}(1/T)}$ [K] | Reference | Type | Note |
|---|---|---|---|---|---|
| 1-isocyanato-3-(trifluoromethyl)-benzene | $4.8\times10^{-3}$ | | Zhang et al. (2010) | Q | 287, 288 |
| $C_8H_4F_3NO$ | 2.5 | | Zhang et al. (2010) | Q | 287, 289 |
| [329-01-1] | $1.3\times10^{-3}$ | | Zhang et al. (2010) | Q | 287, 290 |
| SXJYSIBLFGQAND-UHFFFAOYSA-N | $6.4\times10^{-2}$ | | Zhang et al. (2010) | Q | 287, 291 |
| flonicamid | $2.4\times10^{7}$ | | HSDB (2015) | V | |
| $C_9H_6F_3N_3O$ | $2.4\times10^{7}$ | | Maniere et al. (2011) | ? | 241, 165 |
| [158062-67-0] | | | | | |
| RLQJEEJISHYWON-UHFFFAOYSA-N | | | | | |
| trifluridine | $1.0\times10^{11}$ | | HSDB (2015) | Q | 99 |
| $C_{10}H_{11}F_3N_2O_5$ | | | | | |
| [70-00-8] | | | | | |
| VSQQQLOSPVPRAZ-RRKCRQDMSA-N | | | | | |
| N-(4-amino-2-hydroxyphenyl)-2,2,3,3,4,4,4-heptafluorobutanamide | $2.0\times10^{8}$ | | Zhang et al. (2010) | Q | 287, 288 |
| $C_{10}H_7F_7N_2O_2$ | $2.3\times10^{7}$ | | Zhang et al. (2010) | Q | 287, 289 |
| [847-51-8] | $1.5\times10^{5}$ | | Zhang et al. (2010) | Q | 287, 290 |
| STPOJASQXPXVMS-UHFFFAOYSA-N | $5.7\times10^{6}$ | | Zhang et al. (2010) | Q | 287, 291 |
| fluometuron | $5.8\times10^{3}$ | | Mackay et al. (2006d) | V | |
| $C_{10}H_{11}F_3N_2O$ | $3.8\times10^{3}$ | | HSDB (2015) | C | |
| [2164-17-2] | | | | | |
| RZILCCPWPBTYDO-UHFFFAOYSA-N | | | | | |
| dinitramine | 7.1 | | HSDB (2015) | V | |
| $C_{11}H_{13}F_3N_4O_4$ | 6.5 | | Mackay et al. (2006d) | V | |
| [29091-05-2] | 6.2 | | Suntio et al. (1988) | V | 12 |
| OFDYMSKSGFSLLM-UHFFFAOYSA-N | | | | | |
| 5-methyl-N-[4-(trifluoromethyl)phenyl]-4-isoxazolecarboxamide $C_{12}H_9F_3N_2O_2$ (leflunomide) [75706-12-6] VHOGYURTWQBHIL-UHFFFAOYSA-N | $8.0\times10^{4}$ | | HSDB (2015) | Q | 99 |
| fludioxonil | $1.9\times10^{4}$ | | Duchowicz et al. (2020) | V | 186 |
| $C_{12}H_6F_2N_2O_2$ | $1.6\times10^{4}$ | | Duchowicz et al. (2020) | Q | |
| [131341-86-1] | $1.9\times10^{4}$ | | Maniere et al. (2011) | ? | 241, 165 |
| MUJOIMFVNIBMKC-UHFFFAOYSA-N | | | | | |
| fluconazole | $9.9\times10^{7}$ | | HSDB (2015) | Q | 99 |
| $C_{13}H_{12}F_2N_6O$ | | | | | |
| [86386-73-4] | | | | | |
| RFHAOTPXVQNOHP-UHFFFAOYSA-N | | | | | |



Table A5.1: Organic fluorine (. . . continued)

| Substance<br>Formula<br>(Trivial Name)<br>[CAS Registry Number]<br>InChIKey | $H_s^{cp}$<br>(at $T^\ominus$)<br>$\left[\dfrac{\text{mol}}{\text{m}^3\,\text{Pa}}\right]$ | $\dfrac{\text{d}\ln H_s^{cp}}{\text{d}(1/T)}$<br><br>[K] | Reference | Type | Note |
|---|---|---|---|---|---|
| ethalfluralin<br>$C_{13}H_{14}F_3N_3O_4$<br>[55283-68-6]<br>PTFJIKYUEPWBMS-UHFFFAOYSA-N | $7.6\times10^{-2}$ | | HSDB (2015) | V | |
| benfluralin<br>$C_{13}H_{16}F_3N_3O_4$<br>(benefin)<br>[1861-40-1]<br>SMDHCQAYESWHAE-UHFFFAOYSA-N | $3.4\times10^{-2}$<br><br>$7.5\times10^{-1}$<br>$1.1\times10^{-1}$ | | HSDB (2015)<br>Mackay et al. (2006d)<br>Suntio et al. (1988)<br>Maniere et al. (2011) | V<br>V<br>V<br>? | <br>558<br>12<br>12, 165 |
| trifluralin<br>$C_{13}H_{16}F_3N_3O_4$<br>[1582-09-8]<br>ZSDSQXJSNMTJDA-UHFFFAOYSA-N | $9.5\times10^{-2}$<br>$9.1\times10^{-1}$<br>$1.9\times10^{-1}$<br>$1.7\times10^{-1}$<br><br>$2.5\times10^{-1}$<br>3.8<br>$2.5\times10^{-3}$<br>$9.6\times10^{-2}$<br>$8.3\times10^{-4}$<br>1.7<br>$2.6\times10^{-1}$ | <br><br><br><br><br><br><br><br><br><br><br><br>5000<br>2100 | Rice et al. (1997b)<br>Watanabe (1993)<br>Fendinger et al. (1989)<br>Fendinger et al. (1989)<br>Mackay et al. (2006d)<br>Suntio et al. (1988)<br>Sanders and Seiber (1983)<br>Barcelo and Hennion (1997)<br>HSDB (2015)<br>Goodarzi et al. (2010)<br>Hilal et al. (2008)<br>Modarresi et al. (2007)<br>Kühne et al. (2005)<br>Kühne et al. (2005) | M<br>M<br>M<br>M<br>V<br>V<br>V<br>X<br>C<br>Q<br>Q<br>Q<br>Q<br>? | 12<br><br>72<br>645<br>558<br>12<br>87<br>567<br><br>568<br><br>67<br><br> |
| prodiamine<br>$C_{13}H_{17}F_3N_4O_4$<br>[29091-21-2]<br>RSVPPPHXAASNOL-UHFFFAOYSA-N | 1.3 | | Ebert et al. (2023) | ? | 316 |
| fluorodifen<br>$C_{13}H_7F_3N_2O_5$<br>[15457-05-3]<br>HHMCAJWVGYGUEF-UHFFFAOYSA-N | <br>$6.5\times10^2$ | | Mackay et al. (2006d)<br>MacBean (2012a) | V<br>? | 558 |
| profluralin<br>$C_{14}H_{16}F_3N_3O_4$<br>[26399-36-0]<br>ITVQAKZNYJEWKS-UHFFFAOYSA-N | $3.4\times10^{-2}$<br>$3.2\times10^{-2}$<br>$2.6\times10^{-2}$<br>$3.4\times10^{-2}$ | | HSDB (2015)<br>Mackay et al. (2006d)<br>Suntio et al. (1988)<br>MacBean (2012a) | V<br>V<br>V<br>? | <br><br>12<br> |
| flumequine<br>$C_{14}H_{12}FNO_3$<br>[42835-25-6]<br>DPSPPJIUMHPXMA-UHFFFAOYSA-N | $3.7\times10^7$ | | HSDB (2015) | Q | 99 |
| fluazifop<br>$C_{15}H_{12}F_3NO_4$<br>[69335-91-7]<br>YUVKUEAFAVKILW-UHFFFAOYSA-N | $3.4\times10^7$ | | Ebert et al. (2023) | ? | 316 |





Table A5.1: Organic fluorine (…continued)

| Substance<br>Formula<br>(Trivial Name)<br>[CAS Registry Number]<br>InChIKey | $H_s^{cp}$ (at $T^{\ominus}$) $\left[\dfrac{\mathrm{mol}}{\mathrm{m^3\,Pa}}\right]$ | $\dfrac{\mathrm{d}\ln H_s^{cp}}{\mathrm{d}(1/T)}$ [K] | Reference | Type | Note |
|---|---|---|---|---|---|
| fluazifop-p<br>$C_{15}H_{12}F_3NO_4$<br>[83066-88-0]<br>YUVKUEAFAVKILW-SECBINFHSA-N | $1.0\times10^6$ | | Ebert et al. (2023) | ? | 316 |
| prosulfuron<br>$C_{15}H_{16}F_3N_5O_4S$<br>[94125-34-5]<br>LTUNNEGNEKBSEH-UHFFFAOYSA-N | $>3.3\times10^3$ | | Maniere et al. (2011) | ? | 165 |
| flurprimidol<br>$C_{15}H_{15}N_2O_2F_3$<br>[56425-91-3]<br>VEVZCONIUDBCDC-UHFFFAOYSA-N | $7.5\times10^3$<br>$2.5\times10^4$ | | Duchowicz et al. (2020)<br>Duchowicz et al. (2020) | V<br>Q | 186 |
| flutriafol<br>$C_{16}H_{13}F_2N_3O$<br>[76674-21-0]<br>JWUCHKBSVLQQCO-UHFFFAOYSA-N | $6.1\times10^7$<br>$7.6\times10^4$<br>$7.9\times10^5$ | | Duchowicz et al. (2020)<br>Duchowicz et al. (2020)<br>Maniere et al. (2011) | V<br>Q<br>? | 186<br><br>12, 165 |
| flunitrazepam<br>$C_{16}H_{12}FN_3O_3$<br>[1622-62-4]<br>PPTYJKAXVCCBDU-UHFFFAOYSA-N | $4.3\times10^5$ | | HSDB (2015) | Q | 99 |
| cyhalofop<br>$C_{16}H_{12}FNO_4$<br>[122008-78-0]<br>ROBSGBGTWRRYSK-SNVBAGLBSA-N | $1.7\times10^5$ | | Ebert et al. (2023) | ? | 318 |
| benzpyrimoxan<br>$C_{16}H_{15}F_3N_2O_3$<br>[1449021-97-9]<br>ZYXYTGQFPZEUFX-UHFFFAOYSA-N | $1.1\times10^3$ | | Ebert et al. (2023) | ? | 318 |
| tolprocarb<br>$C_{16}H_{21}F_3N_2O_3$<br>[911499-62-2]<br>RSOBJVBYZCMJOS-CYBMUJFWSA-N | $6.5\times10^4$ | | Ebert et al. (2023) | ? | 318 |
| flutolanil<br>$C_{17}H_{16}F_3NO_2$<br>[66332-96-5]<br>PTCGDEVVHUXTMP-UHFFFAOYSA-N | $3.1\times10^3$<br>$2.5$<br>$6.1\times10^4$ | | Duchowicz et al. (2020)<br>Duchowicz et al. (2020)<br>Maniere et al. (2011) | V<br>Q<br>? | 186<br><br>241, 165 |
| beflubutamid<br>$C_{18}H_{17}F_4NO_2$<br>[113614-08-7]<br>FFQPZWRNXKPNPX-UHFFFAOYSA-N | $9.1\times10^3$ | | Maniere et al. (2011) | ? | 241, 165 |



Table A5.1: Organic fluorine (...continued)

| Substance Formula (Trivial Name) [CAS Registry Number] InChIKey | $H_s^{cp}$ (at $T^{\ominus}$) $\left[\dfrac{\mathrm{mol}}{\mathrm{m^3\,Pa}}\right]$ | $\dfrac{\mathrm{d}\ln H_s^{cp}}{\mathrm{d}(1/T)}$ [K] | Reference | Type | Note |
|---|---|---|---|---|---|
| fluxapyroxad $C_{18}H_{12}F_5N_3O$ [907204-31-3] SXSGXWCSHSVPGB-UHFFFAOYSA-N | $3.3\times10^6$ | | Maniere et al. (2011) | ? | 12, 165 |
| flurtamone $C_{18}H_{14}F_3NO_2$ [96525-23-4] NYRMIJKDBAQCHC-UHFFFAOYSA-N | $1.6\times10^8$ | | Ebert et al. (2023) | ? | 318 |
| picoxystrobin $C_{18}H_{16}F_3NO_4$ [117428-22-5] IBSNKSODLGJUMQ-SDNWHVSQSA-N | $1.4\times10^3$ | | Ebert et al. (2023) | ? | 316 |
| sedaxane $C_{18}H_{19}F_2N_3O$ [874967-67-6] XQJQCBDIXRIYRP-UHFFFAOYSA-N | $2.5\times10^5$ | | Maniere et al. (2011) | ? | 165 |
| penflufen $C_{18}H_{24}FN_3O$ [494793-67-8] GOFJDXZZHFNFLV-UHFFFAOYSA-N | $3.2\times10^4$ | | Ebert et al. (2023) | ? | 318 |
| fluazifop-p-butyl $C_{19}H_{20}F_3NO_4$ [79241-46-6] VAIZTNZGPYBOGF-CYBMUJFWSA-N | $2.0\times10^1$ | | Maniere et al. (2011) | ? | 12, 165 |
| picolinafen $C_{19}H_{12}F_4N_2O_2$ [137641-05-5] CWKFPEBMTGKLKX-UHFFFAOYSA-N | $6.2\times10^2$ | | Maniere et al. (2011) | ? | 12, 165 |
| diflufenican $C_{19}H_{11}F_5N_2O_2$ [83164-33-4] WYEHFWKAOXOVJD-UHFFFAOYSA-N | $6.4\times10^1$ $1.1\times10^3$ $3.0\times10^1$ $<8.5\times10^1$ | | Keshavarz et al. (2022) Duchowicz et al. (2020) Duchowicz et al. (2020) Maniere et al. (2011) | Q Q ? ? | 185, 21 241, 165 |
| fluazifop-butyl $C_{19}H_{20}F_3NO_4$ [69806-50-4] VAIZTNZGPYBOGF-UHFFFAOYSA-N | $4.7\times10^1$ | | HSDB (2015) | V | |
| flumioxazin $C_{19}H_{15}FN_2O_4$ [103361-09-7] FOUWCSDKDDHKQP-UHFFFAOYSA-N | $1.6\times10^1$ $1.6\times10^1$ | | HSDB (2015) Maniere et al. (2011) | V ? | 12, 165 |



Table A5.1: Organic fluorine (...continued)

| Substance<br>Formula<br>(Trivial Name)<br>[CAS Registry Number]<br>InChIKey | $H_s^{cp}$<br>(at $T^\ominus$)<br>$\left[\dfrac{\text{mol}}{\text{m}^3\,\text{Pa}}\right]$ | $\dfrac{\text{d}\ln H_s^{cp}}{\text{d}(1/T)}$<br><br>[K] | Reference | Type | Note |
|---|---|---|---|---|---|
| fluridone<br>$C_{19}H_{14}F_3NO$<br>[59756-60-4]<br>YWBVHLJPRPCRSD-UHFFFAOYSA-N | $1.2\times10^3$<br>$2.8\times10^3$<br><br>$1.9\times10^2$ | | Duchowicz et al. (2020)<br>HSDB (2015)<br>Mackay et al. (2006d)<br>Duchowicz et al. (2020) | V<br>V<br>V<br>Q | 186<br><br>558 |
| cyhalofop-butyl<br>$C_{20}H_{20}FNO_4$<br>[122008-85-9]<br>TYIYMOAHACZAMQ-CQSZACIVSA-N | $1.0\times10^3$<br>$1.1\times10^3$ | | MacBean (2012b)<br>Maniere et al. (2011) | X<br>? | 350<br>241, 165 |
| raltegravir<br>$C_{20}H_{21}FN_6O_5$<br>[518048-05-0]<br>CZFFBEXEKNGXKS-UHFFFAOYSA-N | $1.1\times10^{17}$ | | HSDB (2015) | Q | 99 |
| fluacrypyrim<br>$C_{20}H_{21}N_2O_5F_3$<br>[229977-93-9]<br>MXWAGQASUDSFBG-RVDMUPIBSA-N | $3.0\times10^2$ | | MacBean (2012a) | ? | 12 |
| trifloxystrobin<br>$C_{20}H_{19}F_3N_2O_4$<br>[141517-21-7]<br>ONCZDRURRATYFI-UHFFFAOYSA-N | $4.3\times10^2$<br>$4.3\times10^2$ | | MacBean (2012b)<br>Maniere et al. (2011) | X<br>? | 350<br>165 |
| isopyrazam<br>$C_{20}H_{23}F_2N_3O$<br>[881685-58-1]<br>XTDZGXBTXBEZDN-UHFFFAOYSA-N | $5.3\times10^3$<br>$2.7\times10^4$ | | Maniere et al. (2011)<br>Maniere et al. (2011) | ?<br>? | 241, 165<br>241, 165 |
| syn-isopyrazam<br>$C_{20}H_{23}F_2N_3O$<br>[683777-13-1]<br>XTDZGXBTXBEZDN-HEHGZKQESA-N | $5.2\times10^6$ | | Ebert et al. (2023) | ? | 318 |
| anti-isopyrazam<br>$C_{20}H_{23}F_2N_3O$<br>[683777-14-2]<br>XTDZGXBTXBEZDN-XEZPLFJOSA-N | $2.6\times10^6$ | | Ebert et al. (2023) | ? | 318 |
| etoxazole<br>$C_{21}H_{23}F_2NO_2$<br>[153233-91-1]<br>IXSZQYVWNJNRAL-UHFFFAOYSA-N | $9.6\times10^1$<br>$9.9\times10^1$<br>$1.7\times10^2$<br>$2.8\times10^1$ | | Duchowicz et al. (2020)<br>HSDB (2015)<br>Duchowicz et al. (2020)<br>Maniere et al. (2011) | V<br>V<br>Q<br>? | 186<br><br><br>165 |
| droperidol<br>$C_{22}H_{22}FN_3O_2$<br>[548-73-2]<br>RMEDXOLNCUSCGS-UHFFFAOYSA-N | $3.7\times10^{11}$ | | HSDB (2015) | Q | 99 |



Table A5.1: Organic fluorine (...continued)

| Substance<br>Formula<br>(Trivial Name)<br>[CAS Registry Number]<br>InChIKey | $H_s^{cp}$ (at $T^\ominus$) $\left[\dfrac{\text{mol}}{\text{m}^3\,\text{Pa}}\right]$ | $\dfrac{\text{d}\ln H_s^{cp}}{\text{d}(1/T)}$ [K] | Reference | Type | Note |
|---|---|---|---|---|---|
| paliperidone<br>$C_{23}H_{27}FN_4O_3$<br>[144598-75-4]<br>PMXMIIMHBWHSKN-UHFFFAOYSA-N | $1.2\times10^{15}$ | | HSDB (2015) | Q | 99 |
| risperidone<br>$C_{23}H_{27}FN_4O_2$<br>[106266-06-2]<br>RAPZEAPATHNIPO-UHFFFAOYSA-N | $4.5\times10^{10}$ | | HSDB (2015) | Q | 99 |
| ezetimibe<br>$C_{24}H_{21}F_2NO_3$<br>[163222-33-1]<br>OLNTVTPDXPETLC-XPWALMASSA-N | $2.2\times10^{12}$ | | HSDB (2015) | Q | 99 |
| cyflumetofen<br>$C_{24}H_{24}F_3NO_4$<br>[400882-07-7]<br>AWSZRJQNBMEZOI-UHFFFAOYSA-N | $>1.1\times10^{1}$ | | Maniere et al. (2011) | ? | 241, 165 |
| acrinathrin<br>$C_{26}H_{21}F_6NO_5$<br>[101007-06-1]<br>YLFSVIMMRPNPFK-WEQBUNFVSA-N | $9.3\times10^{1}$ | | Maniere et al. (2011) | ? | 12, 165 |
| cerivastatin<br>$C_{26}H_{34}FNO_5$<br>[145599-86-6]<br>SEERZIQQUAZTOL-ANMDKAQQSA-N | $1.7\times10^{13}$ | | HSDB (2015) | Q | 99 |
| flucythrinate, isomer 1<br>$C_{26}H_{23}F_2NO_4$<br>[70124-77-5]<br>GBIHOLCMZGAKNG-UHFFFAOYSA-N | $1.1\times10^{2}$<br>$9.3\times10^{2}$ | | HSDB (2015)<br>Mackay et al. (2006d) | V<br>V | |
| PFBHA-methanal<br>$H_2C{=}NOCH_2C_6F_5$<br>[86356-73-2]<br>SRTQFRQWTUMMTC-UHFFFAOYSA-N | $1.6\times10^{-2}$ | 7200 | Destaillats and Charles (2002) | M | |
| PFBHA-ethanal<br>$CH_3CH{=}NOCH_2C_6F_5$<br>[114611-59-5]<br>AKDRYEADQPNLOH-UHFFFAOYSA-N | $1.9\times10^{-2}$ | 5400 | Destaillats and Charles (2002) | M | |
| PFBHA-propanone<br>$(CH_3)_2C{=}NOCH_2C_6F_5$<br>[899828-53-6]<br>DLIFNTQMBOCKTL-UHFFFAOYSA-N | $1.1\times10^{-2}$ | 3800 | Destaillats and Charles (2002) | M | |



Table A5.1: Organic fluorine (...continued)

| Substance<br>Formula<br>(Trivial Name)<br>[CAS Registry Number]<br>InChIKey | $H_s^{cp}$<br>(at $T^{\ominus}$)<br>$\left[\dfrac{\text{mol}}{\text{m}^3\,\text{Pa}}\right]$ | $\dfrac{\text{d}\ln H_s^{cp}}{\text{d}(1/T)}$<br><br>[K] | Reference | Type | Note |
|---|---|---|---|---|---|
| PFBHA-butanone<br>$(C_2H_5)(CH_3)C{=}NOCH_2C_6F_5$<br>LNDQFOSWZJYEIC-UHFFFAOYSA-N | $4.7{\times}10^{-3}$ | 6000 | Destaillats and Charles (2002) | M | |
| PFBHA-2-pentanone<br>$(C_3H_7)(CH_3)C{=}NOCH_2C_6F_5$<br>YKQOTQPNOJSJFA-UHFFFAOYSA-N | $3.7{\times}10^{-3}$ | 2200 | Destaillats and Charles (2002) | M | |
| PFBHA-hexanal<br>$C_5H_{11}CH{=}NOCH_2C_6F_5$<br>GPAVMFOMYSGJDM-UHFFFAOYSA-N | $5.8{\times}10^{-3}$ | | Destaillats and Charles (2002) | M | |
| PFBHA-octanal<br>$C_7H_{15}CH{=}NOCH_2C_6F_5$<br>RLSQIXNUITTXKZ-UHFFFAOYSA-N | $7.9{\times}10^{-3}$ | | Destaillats and Charles (2002) | M | |
| PFBHA-decanal<br>$C_9H_{19}CH{=}NOCH_2C_6F_5$<br>NUGDFCWVLOJWOP-UHFFFAOYSA-N | $2.4{\times}10^{-2}$ | | Destaillats and Charles (2002) | M | |
| PFBHA-propenal<br>$CH_2CHCH{=}NOCH_2C_6F_5$<br>ICDUEGOPUWNJNF-UHFFFAOYSA-N | $9.5{\times}10^{-3}$ | 5400 | Destaillats and Charles (2002) | M | |
| (E)-PFBHA-propenal<br>$C_{10}H_6F_5NO$<br>[932710-55-9]<br>ICDUEGOPUWNJNF-HQYXKAPLSA-N | $9.5{\times}10^{-3}$ | | Ebert et al. (2023) | ? | 579 |
| PFBHA-crotonaldehyde<br>$CH_3CHCHCH{=}NOCH_2C_6F_5$<br>[932710-52-6]<br>QNPFFCQTVXPCLD-UHFFFAOYSA-N | $6.8{\times}10^{-3}$ | 3400 | Destaillats and Charles (2002) | M | |
| PFBHA-benzaldehyde<br>$C_6H_5CH{=}NOCH_2C_6F_5$<br>UKSAZCDAGVHMRF-UHFFFAOYSA-N | $5.0{\times}10^{-3}$ | 2000 | Destaillats and Charles (2002) | M | |
| PFBHA-4-methyl-benzaldehyde<br>$C_8H_8{=}NOCH_2C_6F_5$<br>UCASBURGYXMQLW-UHFFFAOYSA-N | $6.6{\times}10^{-3}$ | | Destaillats and Charles (2002) | M | |
| PFBHA-9-fluorenone<br>$C_{13}H_8{=}NOCH_2C_6F_5$<br>UABDVZDYKDIZFO-UHFFFAOYSA-N | $1.1{\times}10^{-2}$ | | Destaillats and Charles (2002) | M | |
| PFBHA-ethanedial<br>$(HC{=}NOCH_2C_6F_5)_2$<br>[618858-54-1]<br>VNVBOBJSRGZDCW-UHFFFAOYSA-N | $1.6{\times}10^{-2}$ | | Destaillats and Charles (2002) | M | |



Table A5.1: Organic fluorine (...continued)

| Substance<br>Formula<br>(Trivial Name)<br>[CAS Registry Number]<br>InChIKey | $H_s^{cp}$<br>(at $T^{\ominus}$)<br><br>$\left[\dfrac{\text{mol}}{\text{m}^3\,\text{Pa}}\right]$ | $\dfrac{\text{d}\ln H_s^{cp}}{\text{d}(1/T)}$<br><br>[K] | Reference | Type | Note |
|---|---|---|---|---|---|
| PFBHA-1-hydroxypropanone<br>$(CH_2OH)(CH_3)C{=}NOCH_2C_6F_5$<br>BCTASVPUYLXFBW-UHFFFAOYSA-N | $2.7{\times}10^{-2}$ | | Destaillats and Charles (2002) | M | |
| PFBHA-3-hydroxy-3-methyl-2-butanone<br>$(HOC_3H_6)(CH_3)C{=}NOCH_2C_6F_5$<br>VNWPJWLGZUDTBZ-UHFFFAOYSA-N | $1.2{\times}10^{-2}$ | | Destaillats and Charles (2002) | M | |



## A6   Organic species with chlorine (Cl)

### A6.1   Chlorocarbons (C, H, Cl)

Table A6.1: Chlorocarbons (C, H, Cl)

| Substance Formula (Trivial Name) [CAS Registry Number] InChIKey | $H_s^{cp}$ (at $T^{\ominus}$) $\left[\dfrac{\text{mol}}{\text{m}^3\,\text{Pa}}\right]$ | $\dfrac{\mathrm{d}\ln H_s^{cp}}{\mathrm{d}(1/T)}$ [K] | Reference | Type | Note |
|---|---|---|---|---|---|
| chloromethane | $1.0\times10^{-3}$ | 2900 | Schwardt et al. (2021) | L | 1 |
| CH$_3$Cl | $1.0\times10^{-3}$ | 2900 | Burkholder et al. (2019) | L | 1 |
| (methyl chloride) | $8.7\times10^{-4}$ | 3400 | Burkholder et al. (2019) | L | 70 |
| [74-87-3] | $1.1\times10^{-3}$ | 3300 | Burkholder et al. (2015) | L | |
| NEHMKBQYUWJMIP-UHFFFAOYSA-N | $8.7\times10^{-4}$ | 3400 | Burkholder et al. (2015) | L | 70 |
| | $1.0\times10^{-3}$ | 2800 | Brockbank (2013) | L | 1 |
| | $1.3\times10^{-3}$ | 3300 | Sander et al. (2011) | L | 646 |
| | $1.1\times10^{-3}$ | 3300 | Warneck (2007) | L | |
| | $1.3\times10^{-3}$ | 3300 | Sander et al. (2006) | L | 647 |
| | $1.1\times10^{-3}$ | 3300 | Staudinger and Roberts (2001) | L | |
| | $1.1\times10^{-3}$ | | Mackay and Shiu (1981) | L | |
| | $1.0\times10^{-3}$ | 2800 | Wilhelm et al. (1977) | L | |
| | $7.9\times10^{-4}$ | 2400 | Hiatt (2013) | M | |
| | $9.1\times10^{-4}$ | 2000 | Chen et al. (2012) | M | |
| | $8.8\times10^{-4}$ | 3200 | Moore (2000) | M | 70 |
| | $9.3\times10^{-4}$ | 3300 | Moore et al. (1995) | M | 70 |
| | $8.5\times10^{-4}$ | 2800 | Reichl (1995) | M | 648 |
| | $1.1\times10^{-3}$ | 3000 | Elliott and Rowland (1993) | M | |
| | $1.2\times10^{-3}$ | 4200 | Gossett (1987) | M | |
| | $1.4\times10^{-3}$ | | Pearson and McConnell (1975) | M | 649, 12 |
| | $1.1\times10^{-3}$ | 2600 | Swain and Thornton (1962) | M | |
| | $9.9\times10^{-4}$ | 2500 | Boggs and Buck (1958) | M | |
| | $1.0\times10^{-3}$ | 2900 | Glew and Moelwyn-Hughes (1953) | M | 650 |
| | $1.0\times10^{-3}$ | | Mackay et al. (2006b) | V | |
| | $4.2\times10^{-4}$ | | Lide and Frederikse (1995) | V | |
| | $1.0\times10^{-3}$ | | Mackay et al. (1993) | V | |
| | $1.1\times10^{-3}$ | | Dilling (1977) | V | 651 |
| | $1.2\times10^{-3}$ | | Dilling (1977) | V | 12 |
| | $9.9\times10^{-4}$ | | Hine and Mookerjee (1975) | V | |
| | $1.2\times10^{-3}$ | | Yaws (2003) | X | 237 |
| | $2.9\times10^{-4}$ | -630 | Goldstein (1982) | X | 298 |
| | $2.5\times10^{-5}$ | | Ryan et al. (1988) | C | |
| | $1.1\times10^{-3}$ | | Hayer et al. (2022) | Q | 20 |
| | $2.7\times10^{-4}$ | | Wang et al. (2017) | Q | 80, 238 |
| | $1.4\times10^{-3}$ | | Wang et al. (2017) | Q | 80, 239 |
| | $1.8\times10^{-3}$ | | Wang et al. (2017) | Q | 80, 240 |
| | $9.9\times10^{-4}$ | | Li et al. (2014) | Q | 241 |
| | $6.8\times10^{-4}$ | | Gharagheizi et al. (2012) | Q | |
| | $7.8\times10^{-4}$ | | Raventos-Duran et al. (2010) | Q | 271, 243 |
| | $1.2\times10^{-3}$ | | Raventos-Duran et al. (2010) | Q | 244 |
| | $1.2\times10^{-3}$ | | Raventos-Duran et al. (2010) | Q | 245 |
| | $1.0\times10^{-3}$ | | Gharagheizi et al. (2010) | Q | 246 |
| | $1.0\times10^{-3}$ | | Hilal et al. (2008) | Q | |





Table A6.1: Chlorocarbons (C, H, Cl) (... continued)

| Substance Formula (Trivial Name) [CAS Registry Number] InChIKey | $H_s^{cp}$ (at $T^{\ominus}$) $\left[\dfrac{\mathrm{mol}}{\mathrm{m^3\,Pa}}\right]$ | $\dfrac{\mathrm{d}\ln H_s^{cp}}{\mathrm{d}(1/T)}$ [K] | Reference | Type | Note |
|---|---|---|---|---|---|
| | $1.9\times10^{-3}$ | | Modarresi et al. (2007) | Q | 67 |
| | | 2600 | Kühne et al. (2005) | Q | |
| | $1.1\times10^{-3}$ | | Yaffe et al. (2003) | Q | 248, 249 |
| | $8.6\times10^{-4}$ | | Yao et al. (2002) | Q | 229 |
| | $1.0\times10^{-3}$ | | English and Carroll (2001) | Q | 230, 231 |
| | $3.7\times10^{-4}$ | | Katritzky et al. (1998) | Q | |
| | $8.6\times10^{-4}$ | | Suzuki et al. (1992) | Q | 232 |
| | $3.9\times10^{-4}$ | | Nirmalakhandan and Speece (1988) | Q | |
| | $8.6\times10^{-4}$ | | Irmann (1965) | Q | |
| | $1.1\times10^{-3}$ | | Mackay et al. (2006b) | ? | |
| | | 2700 | Kühne et al. (2005) | ? | |
| | $1.2\times10^{-3}$ | | Yaws (1999) | ? | 21 |
| | $6.9\times10^{-4}$ | | Abraham and Weathersby (1994) | ? | 21 |
| | $1.2\times10^{-3}$ | | Yaws and Yang (1992) | ? | 21 |
| | $1.0\times10^{-3}$ | | Abraham et al. (1990) | ? | |
| dichloromethane $CH_2Cl_2$ (methylene chloride) [75-09-2] YMWUJEATGCHHMB-UHFFFAOYSA-N | $3.9\times10^{-3}$ | 3500 | Schwardt et al. (2021) | L | 1 |
| | $3.9\times10^{-3}$ | 3700 | Burkholder et al. (2019) | L | |
| | $3.5\times10^{-3}$ | 3900 | Burkholder et al. (2019) | L | 70 |
| | $3.9\times10^{-3}$ | 3700 | Burkholder et al. (2015) | L | |
| | $3.5\times10^{-3}$ | 3900 | Burkholder et al. (2015) | L | 70 |
| | $3.7\times10^{-3}$ | 3300 | Brockbank (2013) | L | 1 |
| | $3.6\times10^{-3}$ | 4100 | Sander et al. (2011) | L | |
| | $3.9\times10^{-3}$ | 3700 | Warneck (2007) | L | |
| | $3.6\times10^{-3}$ | 4100 | Sander et al. (2006) | L | |
| | $3.6\times10^{-3}$ | 4100 | Staudinger and Roberts (2001) | L | |
| | $3.6\times10^{-3}$ | 4100 | Staudinger and Roberts (1996) | L | |
| | $3.8\times10^{-3}$ | | Mackay and Shiu (1981) | L | |
| | $4.0\times10^{-3}$ | 3900 | Hiatt (2013) | M | |
| | $3.5\times10^{-3}$ | 2300 | Chen et al. (2012) | M | |
| | $3.6\times10^{-3}$ | 3700 | Ooki and Yokouchi (2011) | M | 70 |
| | $3.2\times10^{-3}$ | | Helburn et al. (2008) | M | |
| | $4.3\times10^{-3}$ | 3500 | Lutsyk et al. (2005) | M | |
| | $3.3\times10^{-3}$ | 4200 | Moore (2000) | M | 70 |
| | $3.9\times10^{-3}$ | | David et al. (2000) | M | 72 |
| | $3.4\times10^{-3}$ | | McIntosh and Heffron (2000) | M | 14 |
| | $4.1\times10^{-3}$ | | Ryu and Park (1999) | M | |
| | $3.4\times10^{-3}$ | | Chiang et al. (1998) | M | 652, 12 |
| | 3.7 | | Welke et al. (1998) | M | |
| | $5.1\times10^{-3}$ | | Hovorka and Dohnal (1997) | M | 12 |
| | $3.7\times10^{-3}$ | 3200 | Kondoh and Nakajima (1997) | M | |
| | $4.3\times10^{-3}$ | 3500 | Park et al. (1997) | M | |
| | $4.1\times10^{-3}$ | | Hoff et al. (1993) | M | |
| | $3.8\times10^{-3}$ | | Li et al. (1993) | M | |
| | $3.9\times10^{-3}$ | 3400 | Wright et al. (1992) | M | 653 |
| | $3.9\times10^{-3}$ | 3500 | Tse et al. (1992) | M | |
| | $4.4\times10^{-3}$ | | Yu (1992) | M | 12 |
| | $3.4\times10^{-3}$ | | Guitart et al. (1989) | M | 14 |



Table A6.1: Chlorocarbons (C, H, Cl) (. . . continued)

| Substance Formula (Trivial Name) [CAS Registry Number] InChIKey | $H_s^{cp}$ (at $T^\ominus$) $\left[\dfrac{\mathrm{mol}}{\mathrm{m^3\,Pa}}\right]$ | $\dfrac{\mathrm{d}\ln H_s^{cp}}{\mathrm{d}(1/T)}$ [K] | Reference | Type | Note |
|---|---|---|---|---|---|
| | $3.4\times10^{-3}$ | 4200 | Ashworth et al. (1988) | M | 278 |
| | $4.6\times10^{-3}$ | 3800 | Gossett (1987) | M | |
| | $5.7\times10^{-3}$ | | Hellmann (1987) | M | 87 |
| | $5.2\times10^{-3}$ | | Yurteri et al. (1987) | M | 12 |
| | $3.8\times10^{-3}$ | 4500 | Gossett et al. (1985) | M | |
| | $3.4\times10^{-3}$ | 4200 | Lincoff and Gossett (1984) | M | |
| | $3.0\times10^{-3}$ | 3600 | Leighton and Calo (1981) | M | |
| | $3.1\times10^{-3}$ | | Warner et al. (1980) | M | |
| | $2.8\times10^{-3}$ | | Sato and Nakajima (1979b) | M | 14 |
| | $3.3\times10^{-3}$ | | Pearson and McConnell (1975) | M | 649, 12 |
| | $4.2\times10^{-3}$ | 4400 | Hartkopf and Karger (1973) | M | |
| | $4.1\times10^{-3}$ | 4000 | Rex (1906) | M | |
| | $2.7\times10^{-3}$ | | Mackay et al. (2006b) | V | |
| | $3.5\times10^{-3}$ | 4100 | Fogg and Sangster (2003) | V | |
| | $4.0\times10^{-3}$ | | Park et al. (1997) | V | |
| | $5.9\times10^{-3}$ | | Mackay et al. (1993) | V | |
| | $2.9\times10^{-3}$ | | Hwang et al. (1992) | V | |
| | $3.2\times10^{-3}$ | | Warner et al. (1980) | V | |
| | $4.0\times10^{-3}$ | | Dilling (1977) | V | 651 |
| | $1.2\times10^{-2}$ | | Dilling (1977) | V | 153 |
| | $4.3\times10^{-3}$ | | Hine and Mookerjee (1975) | V | |
| | $4.0\times10^{-3}$ | | Dilling et al. (1975) | V | |
| | $4.0\times10^{-3}$ | | Yaws (2003) | X | 237 |
| | $3.1\times10^{-3}$ | 3600 | Goldstein (1982) | X | 298 |
| | $4.2\times10^{-3}$ | | Harrison et al. (1993) | C | |
| | $3.4\times10^{-3}$ | | Harrison et al. (1993) | C | |
| | $4.7\times10^{-3}$ | | Ryan et al. (1988) | C | |
| | $3.1\times10^{-3}$ | | Shen (1982) | C | |
| | $3.7\times10^{-3}$ | | Dilling (1977) | C | |
| | $3.7\times10^{-3}$ | | Dilling et al. (1975) | C | |
| | $5.6\times10^{-3}$ | | Hayer et al. (2022) | Q | 20 |
| | $8.8\times10^{-3}$ | | Keshavarz et al. (2022) | Q | |
| | $2.6\times10^{-3}$ | | Duchowicz et al. (2020) | Q | |
| | $6.2\times10^{-4}$ | | Wang et al. (2017) | Q | 80, 238 |
| | $1.4\times10^{-2}$ | | Wang et al. (2017) | Q | 80, 239 |
| | $9.3\times10^{-3}$ | | Wang et al. (2017) | Q | 80, 240 |
| | $2.9\times10^{-3}$ | | Gharagheizi et al. (2012) | Q | |
| | $3.1\times10^{-3}$ | | Raventos-Duran et al. (2010) | Q | 242, 243 |
| | $1.2\times10^{-2}$ | | Raventos-Duran et al. (2010) | Q | 244 |
| | $9.9\times10^{-4}$ | | Raventos-Duran et al. (2010) | Q | 245 |
| | $4.1\times10^{-3}$ | | Gharagheizi et al. (2010) | Q | 246 |
| | $9.0\times10^{-3}$ | | Hilal et al. (2008) | Q | |
| | $1.8\times10^{-3}$ | | Modarresi et al. (2007) | Q | 67 |
| | | 3000 | Kühne et al. (2005) | Q | |
| | $3.8\times10^{-3}$ | | Yaffe et al. (2003) | Q | 248, 249 |
| | $3.0\times10^{-3}$ | | Yao et al. (2002) | Q | 229 |
| | $1.8\times10^{-3}$ | | English and Carroll (2001) | Q | 230, 260 |





Table A6.1: Chlorocarbons (C, H, Cl) (... continued)

| Substance<br>Formula<br>(Trivial Name)<br>[CAS Registry Number]<br>InChIKey | $H_s^{cp}$<br>(at $T^{\ominus}$)<br>$\left[\dfrac{\mathrm{mol}}{\mathrm{m}^3\,\mathrm{Pa}}\right]$ | $\dfrac{\mathrm{d}\ln H_s^{cp}}{\mathrm{d}(1/T)}$<br><br>[K] | Reference | Type | Note |
|---|---|---|---|---|---|
| | $6.4\times10^{-4}$ | | Katritzky et al. (1998) | Q | |
| | $2.2\times10^{-3}$ | | Nirmalakhandan and Speece (1988) | Q | |
| | $3.0\times10^{-3}$ | | Duchowicz et al. (2020) | ? | 185, 21 |
| | $3.3\times10^{-3}$ | | Mackay et al. (2006b) | ? | |
| | | 3900 | Kühne et al. (2005) | ? | |
| | $4.0\times10^{-3}$ | | Yaws (1999) | ? | 21 |
| | $2.9\times10^{-3}$ | | Abraham and Weathersby (1994) | ? | 21 |
| | $3.3\times10^{-3}$ | | Mackay et al. (1993) | ? | |
| | $4.0\times10^{-3}$ | | Yaws and Yang (1992) | ? | 21 |
| | $3.7\times10^{-3}$ | | Abraham et al. (1990) | ? | |
| dichloromethane-d2<br>$CD_2Cl_2$<br>(methylene chloride-d2)<br>[1665-00-5]<br>YMWUJEATGCHHMB-DICFDUPASA-N | $3.8\times10^{-3}$ | 4600 | Hiatt (2013) | M | |
| trichloromethane<br>$CHCl_3$<br>(chloroform)<br>[67-66-3]<br>HEDRZPFGACZZDS-UHFFFAOYSA-N | $2.7\times10^{-3}$ | 4200 | Schwardt et al. (2021) | L | 1 |
| | $2.6\times10^{-3}$ | 4300 | Burkholder et al. (2019) | L | |
| | $2.0\times10^{-3}$ | 4400 | Burkholder et al. (2019) | L | 70 |
| | $2.6\times10^{-3}$ | 4300 | Burkholder et al. (2015) | L | |
| | $2.0\times10^{-3}$ | 4400 | Burkholder et al. (2015) | L | 70 |
| | $2.3\times10^{-3}$ | 4200 | Brockbank (2013) | L | 1, 654 |
| | $2.5\times10^{-3}$ | 4500 | Sander et al. (2011) | L | |
| | $2.6\times10^{-3}$ | 4300 | Warneck (2007) | L | |
| | $2.5\times10^{-3}$ | 4500 | Sander et al. (2006) | L | |
| | $2.5\times10^{-3}$ | 4500 | Staudinger and Roberts (2001) | L | |
| | $2.5\times10^{-3}$ | 4500 | Staudinger and Roberts (1996) | L | |
| | $2.6\times10^{-3}$ | | Mackay and Shiu (1981) | L | |
| | $1.6\times10^{-3}$ | | Steward et al. (1973) | L | 14 |
| | $2.6\times10^{-3}$ | 3900 | Allott et al. (1973) | L | |
| | $2.8\times10^{-3}$ | 4500 | Hiatt (2013) | M | |
| | $2.5\times10^{-3}$ | 3900 | Chen et al. (2012) | M | |
| | $3.1\times10^{-3}$ | | Ruiz-Bevia and Fernandez-Torres (2010) | M | |
| | $2.7\times10^{-3}$ | 4500 | Lutsyk et al. (2005) | M | |
| | $1.4\times10^{-3}$ | | Zhang et al. (2002) | M | 14 |
| | $2.3\times10^{-3}$ | 4200 | Görgényi et al. (2002) | M | 655 |
| | $2.0\times10^{-3}$ | 4600 | Moore (2000) | M | 70 |
| | $2.4\times10^{-3}$ | | David et al. (2000) | M | 72 |
| | $2.7\times10^{-3}$ | | Ryu and Park (1999) | M | |
| | $3.0\times10^{-3}$ | | Dohnal and Hovorka (1999) | M | 12 |
| | $3.0\times10^{-3}$ | | Chiang et al. (1998) | M | 12 |
| | $3.4\times10^{-3}$ | | Welke et al. (1998) | M | |
| | $3.2\times10^{-3}$ | | Hovorka and Dohnal (1997) | M | 12 |
| | $2.7\times10^{-3}$ | 3400 | Kondoh and Nakajima (1997) | M | |
| | $2.6\times10^{-3}$ | 3400 | Park et al. (1997) | M | |
| | $2.2\times10^{-3}$ | 4700 | Turner et al. (1996) | M | |



Table A6.1: Chlorocarbons (C, H, Cl) (. . . continued)

| Substance Formula (Trivial Name) [CAS Registry Number] InChIKey | $H_s^{cp}$ (at $T^{\ominus}$) $\left[\dfrac{\mathrm{mol}}{\mathrm{m}^3\,\mathrm{Pa}}\right]$ | $\dfrac{\mathrm{d}\ln H_s^{cp}}{\mathrm{d}(1/T)}$ [K] | Reference | Type | Note |
|---|---|---|---|---|---|
| | $2.2\times10^{-3}$ | 4200 | Moore et al. (1995) | M | 656, 70 |
| | $2.6\times10^{-3}$ | 4400 | Dewulf et al. (1995) | M | |
| | $2.5\times10^{-3}$ | | Hoff et al. (1993) | M | |
| | $2.4\times10^{-3}$ | | Li et al. (1993) | M | |
| | $2.6\times10^{-3}$ | 4000 | Wright et al. (1992) | M | 657 |
| | $4.8\times10^{-3}$ | 7300 | Tancrède and Yanagisawa (1990) | M | |
| | $2.4\times10^{-3}$ | 2000 | Lamarche and Droste (1989) | M | 345 |
| | $2.1\times10^{-3}$ | | Guitart et al. (1989) | M | 14 |
| | $2.3\times10^{-3}$ | 5000 | Ashworth et al. (1988) | M | 278 |
| | $2.7\times10^{-3}$ | 4600 | Gossett (1987) | M | |
| | $2.6\times10^{-3}$ | 4300 | Munz and Roberts (1987) | M | |
| | $2.9\times10^{-3}$ | | Hellmann (1987) | M | 87 |
| | $3.3\times10^{-3}$ | | Munz and Roberts (1986) | M | |
| | $2.5\times10^{-3}$ | 4300 | Gossett et al. (1985) | M | |
| | $2.5\times10^{-3}$ | 5200 | Nicholson et al. (1984) | M | |
| | $2.3\times10^{-3}$ | 4200 | Lincoff and Gossett (1984) | M | |
| | $2.0\times10^{-3}$ | 3900 | Hunter-Smith et al. (1983) | M | 70, 658 |
| | $2.5\times10^{-3}$ | 4000 | Leighton and Calo (1981) | M | |
| | $1.5\times10^{-3}$ | 5600 | Ervin et al. (1980) | M | |
| | $2.9\times10^{-3}$ | | Warner et al. (1980) | M | |
| | $2.4\times10^{-3}$ | 7200 | Balls (1980) | M | |
| | $1.4\times10^{-3}$ | | Sato and Nakajima (1979b) | M | 14 |
| | $3.5\times10^{-3}$ | | Pearson and McConnell (1975) | M | 649, 12 |
| | $2.8\times10^{-3}$ | 5100 | Hartkopf and Karger (1973) | M | |
| | $1.6\times10^{-3}$ | | Bachofen and Farhi (1971) | M | 14 |
| | $2.6\times10^{-3}$ | 4600 | Rex (1906) | M | |
| | $2.6\times10^{-3}$ | | Mackay et al. (2006b) | V | |
| | $2.6\times10^{-3}$ | 4400 | Fogg and Sangster (2003) | V | |
| | $2.5\times10^{-3}$ | | Park et al. (1997) | V | |
| | $2.6\times10^{-3}$ | | Mackay et al. (1993) | V | |
| | $2.6\times10^{-3}$ | | Hwang et al. (1992) | V | |
| | $5.5\times10^{-3}$ | | McLachlan et al. (1990) | V | 373 |
| | $3.1\times10^{-3}$ | | Warner et al. (1980) | V | |
| | $2.8\times10^{-3}$ | | Smith and Bomberger (1980) | V | 24 |
| | $2.5\times10^{-3}$ | | Dilling (1977) | V | 651 |
| | $9.0\times10^{-3}$ | | Dilling (1977) | V | 153 |
| | $2.3\times10^{-3}$ | | Hine and Mookerjee (1975) | V | |
| | $2.5\times10^{-3}$ | | Dilling et al. (1975) | V | |
| | $2.2\times10^{-3}$ | 4700 | Winkler (1906) | V | |
| | $2.5\times10^{-3}$ | 4100 | Barr and Newsham (1987) | X | 298 |
| | $3.0\times10^{-3}$ | 4400 | Goldstein (1982) | X | 298 |
| | $2.4\times10^{-3}$ | | Harrison et al. (1993) | C | |
| | $3.4\times10^{-3}$ | | Harrison et al. (1993) | C | |
| | $3.4\times10^{-3}$ | | Ryan et al. (1988) | C | |
| | $2.7\times10^{-3}$ | | Nicholson et al. (1984) | C | |
| | $2.1\times10^{-3}$ | | Nicholson et al. (1984) | C | 12 |
| | $2.9\times10^{-3}$ | | Shen (1982) | C | |



Table A6.1: Chlorocarbons (C, H, Cl) (. . . continued)

| Substance<br>Formula<br>(Trivial Name)<br>[CAS Registry Number]<br>InChIKey | $H_s^{cp}$<br>(at $T^{\ominus}$)<br>$\left[\dfrac{\text{mol}}{\text{m}^3\,\text{Pa}}\right]$ | $\dfrac{\mathrm{d}\ln H_s^{cp}}{\mathrm{d}(1/T)}$<br><br>[K] | Reference | Type | Note |
|---|---|---|---|---|---|
| | $3.1\times10^{-3}$ | | Dilling (1977) | C | |
| | $3.1\times10^{-3}$ | | Dilling et al. (1975) | C | |
| | $2.6\times10^{-3}$ | | Hayer et al. (2022) | Q | 20 |
| | $1.1\times10^{-3}$ | | Wang et al. (2017) | Q | 80, 238 |
| | $4.3\times10^{-3}$ | | Wang et al. (2017) | Q | 80, 239 |
| | $4.5\times10^{-3}$ | | Wang et al. (2017) | Q | 80, 240 |
| | $3.7\times10^{-3}$ | | Gharagheizi et al. (2012) | Q | |
| | $2.5\times10^{-3}$ | | Raventos-Duran et al. (2010) | Q | 242, 243 |
| | $3.1\times10^{-3}$ | | Raventos-Duran et al. (2010) | Q | 244 |
| | $3.1\times10^{-3}$ | | Raventos-Duran et al. (2010) | Q | 245 |
| | $3.2\times10^{-3}$ | | Hilal et al. (2008) | Q | |
| | $1.3\times10^{-3}$ | | Modarresi et al. (2007) | Q | 67 |
| | | 3300 | Kühne et al. (2005) | Q | |
| | $4.1\times10^{-3}$ | | Yao et al. (2002) | Q | 229 |
| | $1.0\times10^{-3}$ | | English and Carroll (2001) | Q | 230, 274 |
| | $2.6\times10^{-4}$ | | Katritzky et al. (1998) | Q | |
| | $3.9\times10^{-3}$ | | Nirmalakhandan and Speece (1988) | Q | |
| | $2.3\times10^{-3}$ | | Arbuckle (1983) | Q | |
| | $2.3\times10^{-3}$ | | Mackay et al. (2006b) | ? | |
| | | 4300 | Kühne et al. (2005) | ? | |
| | $2.4\times10^{-3}$ | | Yaws (1999) | ? | 21 |
| | $1.6\times10^{-3}$ | | Abraham and Weathersby (1994) | ? | 21 |
| | $2.3\times10^{-3}$ | | Mackay et al. (1993) | ? | |
| | $2.4\times10^{-3}$ | | Yaws and Yang (1992) | ? | 21 |
| | $2.5\times10^{-3}$ | | Abraham et al. (1990) | ? | |
| tetrachloromethane<br>$CCl_4$<br>(carbontetrachloride)<br>[56-23-5]<br>VZGDMQKNWNREIO-UHFFFAOYSA-N | $3.4\times10^{-4}$ | 4200 | Schwardt et al. (2021) | L | 1 |
| | $3.6\times10^{-4}$ | 4300 | Burkholder et al. (2019) | L | |
| | $2.6\times10^{-4}$ | 4200 | Burkholder et al. (2019) | L | 70 |
| | $3.6\times10^{-4}$ | 4300 | Burkholder et al. (2015) | L | |
| | $2.6\times10^{-4}$ | 4200 | Burkholder et al. (2015) | L | 70 |
| | $3.5\times10^{-4}$ | 4200 | Brockbank (2013) | L | 1, 659 |
| | $3.4\times10^{-4}$ | 4200 | Sander et al. (2011) | L | |
| | $3.6\times10^{-4}$ | 4300 | Warneck (2007) | L | |
| | $3.4\times10^{-4}$ | 4200 | Sander et al. (2006) | L | |
| | $3.4\times10^{-4}$ | 4200 | Staudinger and Roberts (2001) | L | |
| | $3.4\times10^{-4}$ | 4200 | Staudinger and Roberts (1996) | L | |
| | $5.0\times10^{-4}$ | | Mackay and Shiu (1981) | L | |
| | $5.0\times10^{-4}$ | 4500 | Hiatt (2013) | M | |
| | $3.0\times10^{-4}$ | 4400 | Chen et al. (2012) | M | |
| | $3.4\times10^{-4}$ | 3800 | Lutsyk et al. (2005) | M | |
| | $3.8\times10^{-4}$ | | Ryu and Park (1999) | M | |
| | $4.0\times10^{-4}$ | | Chiang et al. (1998) | M | 12 |
| | $2.9\times10^{-4}$ | 3700 | Bullister and Wisegarver (1998) | M | 660 |
| | $4.4\times10^{-4}$ | 1900 | Kondoh and Nakajima (1997) | M | |
| | $3.9\times10^{-4}$ | 2600 | Park et al. (1997) | M | |
| | $3.8\times10^{-4}$ | 4400 | Dewulf et al. (1995) | M | |
| | $3.6\times10^{-4}$ | | Hoff et al. (1993) | M | |





Table A6.1: Chlorocarbons (C, H, Cl) (... continued)

| Substance<br>Formula<br>(Trivial Name)<br>[CAS Registry Number]<br>InChIKey | $H_s^{cp}$<br>(at $T^{\ominus}$)<br>$\left[\dfrac{\mathrm{mol}}{\mathrm{m^3\,Pa}}\right]$ | $\dfrac{\mathrm{d}\ln H_s^{cp}}{\mathrm{d}(1/T)}$<br><br>[K] | Reference | Type | Note |
|---|---|---|---|---|---|
| | $3.3\times10^{-4}$ | 3600 | Hansen et al. (1993) | M | 281 |
| | $2.3\times10^{-4}$ | | Li and Carr (1993) | M | |
| | $2.9\times10^{-4}$ | 4500 | Wright et al. (1992) | M | 661 |
| | $3.8\times10^{-4}$ | 3600 | Tse et al. (1992) | M | |
| | $3.4\times10^{-4}$ | 4100 | Tancrède and Yanagisawa (1990) | M | |
| | $2.8\times10^{-4}$ | 5600 | Bissonette et al. (1990) | M | |
| | $3.3\times10^{-4}$ | 4000 | Ashworth et al. (1988) | M | 278 |
| | $3.3\times10^{-4}$ | 4400 | Gossett (1987) | M | |
| | $3.3\times10^{-4}$ | 4300 | Munz and Roberts (1987) | M | |
| | $3.3\times10^{-4}$ | | Hellmann (1987) | M | 87 |
| | $4.3\times10^{-4}$ | | Yurteri et al. (1987) | M | 12 |
| | $4.2\times10^{-4}$ | | Munz and Roberts (1986) | M | |
| | $4.1\times10^{-4}$ | 3200 | Hunter-Smith et al. (1983) | M | 658 |
| | $3.6\times10^{-4}$ | 4400 | Leighton and Calo (1981) | M | |
| | $3.3\times10^{-4}$ | | Warner et al. (1980) | M | |
| | $3.2\times10^{-4}$ | 3300 | Balls (1980) | M | |
| | $9.7\times10^{-5}$ | | Sato and Nakajima (1979b) | M | 14 |
| | $4.5\times10^{-4}$ | | Pearson and McConnell (1975) | M | 649, 12 |
| | $3.7\times10^{-4}$ | 5200 | Hartkopf and Karger (1973) | M | |
| | $3.5\times10^{-4}$ | 4400 | Rex (1906) | M | |
| | $3.4\times10^{-4}$ | | Mackay et al. (2006b) | V | |
| | $3.6\times10^{-4}$ | 4200 | Fogg and Sangster (2003) | V | |
| | $4.3\times10^{-4}$ | | Park et al. (1997) | V | |
| | $3.4\times10^{-4}$ | | Mackay et al. (1993) | V | |
| | $3.4\times10^{-4}$ | | Hwang et al. (1992) | V | |
| | $6.7\times10^{-5}$ | | Ballschmiter and Wittlinger (1991) | V | |
| | $3.5\times10^{-4}$ | | Warner et al. (1980) | V | |
| | $4.6\times10^{-4}$ | | Smith and Bomberger (1980) | V | 24 |
| | $3.4\times10^{-4}$ | | Dilling (1977) | V | |
| | $3.4\times10^{-4}$ | | Hine and Mookerjee (1975) | V | |
| | $2.0\times10^{-4}$ | | Pierotti (1965) | T | |
| | $3.4\times10^{-4}$ | | Yaws (2003) | X | 237 |
| | $3.3\times10^{-4}$ | 1100 | Goldstein (1982) | X | 298 |
| | $3.8\times10^{-4}$ | | Harrison et al. (1993) | C | |
| | $2.1\times10^{-4}$ | | Harrison et al. (1993) | C | |
| | $4.5\times10^{-4}$ | | Ryan et al. (1988) | C | |
| | $3.3\times10^{-4}$ | | Shen (1982) | C | |
| | $4.6\times10^{-4}$ | | Dilling (1977) | C | |
| | $3.7\times10^{-4}$ | | Liss and Slater (1974) | C | |
| | $4.9\times10^{-4}$ | | Hayer et al. (2022) | Q | 20 |
| | $5.4\times10^{-4}$ | | Keshavarz et al. (2022) | Q | |
| | $6.3\times10^{-4}$ | | Duchowicz et al. (2020) | Q | |
| | $3.4\times10^{-4}$ | | Li et al. (2014) | Q | 241 |
| | $3.2\times10^{-3}$ | | Gharagheizi et al. (2012) | Q | |
| | $3.1\times10^{-4}$ | | Gharagheizi et al. (2010) | Q | 246 |
| | $5.4\times10^{-4}$ | | Hilal et al. (2008) | Q | |
| | $2.3\times10^{-4}$ | | Modarresi et al. (2007) | Q | 67 |



Table A6.1: Chlorocarbons (C, H, Cl) (...continued)

| Substance Formula (Trivial Name) [CAS Registry Number] InChIKey | $H_s^{cp}$ (at $T^\ominus$) $\left[\dfrac{\mathrm{mol}}{\mathrm{m^3\,Pa}}\right]$ | $\dfrac{\mathrm{d}\ln H_s^{cp}}{\mathrm{d}(1/T)}$ [K] | Reference | Type | Note |
|---|---|---|---|---|---|
| | | 3700 | Kühne et al. (2005) | Q | |
| | $3.5\times10^{-4}$ | | Yaffe et al. (2003) | Q | 248, 249 |
| | $2.1\times10^{-4}$ | | English and Carroll (2001) | Q | 230, 231 |
| | $3.4\times10^{-5}$ | | Katritzky et al. (1998) | Q | |
| | $3.5\times10^{-4}$ | | Nirmalakhandan and Speece (1988) | Q | |
| | $4.1\times10^{-4}$ | | Arbuckle (1983) | Q | |
| | $3.6\times10^{-4}$ | | Duchowicz et al. (2020) | ? | 185, 21 |
| | $1.2\times10^{-4}$ | | MacBean (2012a) | ? | |
| | $3.3\times10^{-4}$ | | Mackay et al. (2006b) | ? | |
| | | 4300 | Kühne et al. (2005) | ? | |
| | $3.4\times10^{-4}$ | | Yaws (1999) | ? | 21 |
| | $1.0\times10^{-4}$ | | Abraham and Weathersby (1994) | ? | 21 |
| | $3.3\times10^{-4}$ | | Mackay et al. (1993) | ? | |
| | $3.3\times10^{-4}$ | | Yaws and Yang (1992) | ? | 21 |
| | $3.5\times10^{-4}$ | | Abraham et al. (1990) | ? | |
| | $4.3\times10^{-4}$ | | Mackay and Yeun (1983) | ? | |
| | $1.1\times10^{-3}$ | | Chiou et al. (1980) | ? | 79 |
| chloroethane $C_2H_5Cl$ [75-00-3] HRYZWHHZPQKTII-UHFFFAOYSA-N | $8.1\times10^{-4}$ | 3000 | Schwardt et al. (2021) | L | 1 |
| | $8.1\times10^{-4}$ | 2900 | Burkholder et al. (2019) | L | 1 |
| | $8.3\times10^{-4}$ | 2800 | Burkholder et al. (2015) | L | |
| | $8.1\times10^{-4}$ | 2900 | Brockbank (2013) | L | 1 |
| | $8.3\times10^{-4}$ | 2800 | Warneck (2007) | L | |
| | $8.4\times10^{-4}$ | 2900 | Staudinger and Roberts (2001) | L | |
| | $8.3\times10^{-4}$ | 2900 | Staudinger and Roberts (1996) | L | |
| | $5.0\times10^{-3}$ | | Mackay and Shiu (1981) | L | |
| | $4.7\times10^{-4}$ | | Steward et al. (1973) | L | 14 |
| | $7.3\times10^{-4}$ | 3500 | Allott et al. (1973) | L | |
| | $8.5\times10^{-4}$ | 3200 | Hiatt (2013) | M | |
| | $7.6\times10^{-4}$ | 3100 | Chen et al. (2012) | M | |
| | $8.9\times10^{-4}$ | 3200 | Maaßen (1995) | M | 662 |
| | $9.3\times10^{-4}$ | 3300 | Reichl (1995) | M | 663 |
| | $7.9\times10^{-4}$ | 2600 | Ashworth et al. (1988) | M | 278 |
| | $8.8\times10^{-4}$ | 3100 | Gossett (1987) | M | |
| | $5.5\times10^{-3}$ | | Mackay et al. (2006b) | V | |
| | $5.5\times10^{-3}$ | | Mackay et al. (1993) | V | |
| | $5.6\times10^{-4}$ | | Hwang et al. (1992) | V | |
| | $8.8\times10^{-4}$ | | Dilling (1977) | V | |
| | $1.2\times10^{-3}$ | | Hine and Mookerjee (1975) | V | |
| | $1.4\times10^{-3}$ | | Yaws (2003) | X | 237, 12 |
| | $6.8\times10^{-4}$ | 750 | Goldstein (1982) | X | 298 |
| | $6.6\times10^{-4}$ | | Ryan et al. (1988) | C | |
| | $6.3\times10^{-4}$ | | Irmann (1965) | C | |
| | $8.0\times10^{-4}$ | | Hayer et al. (2022) | Q | 20 |
| | $2.2\times10^{-4}$ | | Wang et al. (2017) | Q | 80, 238 |
| | $1.4\times10^{-3}$ | | Wang et al. (2017) | Q | 80, 239 |
| | $1.7\times10^{-3}$ | | Wang et al. (2017) | Q | 80, 240 |
| | $7.9\times10^{-4}$ | | Gharagheizi et al. (2012) | Q | |



Table A6.1: Chlorocarbons (C, H, Cl) (...continued)

| Substance<br>Formula<br>(Trivial Name)<br>[CAS Registry Number]<br>InChIKey | $H_s^{cp}$<br>(at $T^{\ominus}$)<br>$\left[\dfrac{\text{mol}}{\text{m}^3\,\text{Pa}}\right]$ | $\dfrac{\text{d}\ln H_s^{cp}}{\text{d}(1/T)}$<br><br>[K] | Reference | Type | Note |
|---|---|---|---|---|---|
| | $6.2\times10^{-4}$ | | Raventos-Duran et al. (2010) | Q | 271, 243 |
| | $1.2\times10^{-3}$ | | Raventos-Duran et al. (2010) | Q | 244 |
| | $9.9\times10^{-4}$ | | Raventos-Duran et al. (2010) | Q | 245 |
| | $9.3\times10^{-4}$ | | Gharagheizi et al. (2010) | Q | 246 |
| | $1.2\times10^{-3}$ | | Hilal et al. (2008) | Q | |
| | $1.0\times10^{-3}$ | | Modarresi et al. (2007) | Q | 67 |
| | | 3000 | Kühne et al. (2005) | Q | |
| | $1.1\times10^{-3}$ | | Yaffe et al. (2003) | Q | 248, 249 |
| | $4.8\times10^{-4}$ | | Yao et al. (2002) | Q | 229 |
| | $9.7\times10^{-4}$ | | English and Carroll (2001) | Q | 230, 231 |
| | $9.0\times10^{-4}$ | | Katritzky et al. (1998) | Q | |
| | $6.1\times10^{-4}$ | | Suzuki et al. (1992) | Q | 232 |
| | $7.9\times10^{-4}$ | | Nirmalakhandan and Speece (1988) | Q | |
| | $7.6\times10^{-4}$ | | Irmann (1965) | Q | |
| | $9.8\times10^{-4}$ | | Mackay et al. (2006b) | ? | |
| | | 2900 | Kühne et al. (2005) | ? | |
| | $1.4\times10^{-3}$ | | Yaws (1999) | ? | 21, 12 |
| | $4.8\times10^{-4}$ | | Abraham and Weathersby (1994) | ? | 21 |
| | $9.8\times10^{-4}$ | | Mackay et al. (1993) | ? | |
| | $1.4\times10^{-3}$ | | Yaws and Yang (1992) | ? | 21, 12 |
| | $1.2\times10^{-3}$ | | Abraham et al. (1990) | ? | |
| 1,1-dichloroethane<br>$CHCl_2CH_3$<br>[75-34-3]<br>SCYULBFZEHDVBN-UHFFFAOYSA-N | $1.7\times10^{-3}$ | 3900 | Schwardt et al. (2021) | L | 1 |
| | $1.7\times10^{-3}$ | 4000 | Burkholder et al. (2019) | L | 1 |
| | $1.5\times10^{-3}$ | 3900 | Burkholder et al. (2019) | L | 70 |
| | $1.7\times10^{-3}$ | 4100 | Burkholder et al. (2015) | L | |
| | $1.5\times10^{-3}$ | 3900 | Burkholder et al. (2015) | L | 70 |
| | $1.7\times10^{-3}$ | 4000 | Brockbank (2013) | L | 1 |
| | $1.7\times10^{-3}$ | 4100 | Warneck (2007) | L | |
| | $1.8\times10^{-3}$ | 4100 | Fogg and Sangster (2003) | L | |
| | $1.6\times10^{-3}$ | 3700 | Staudinger and Roberts (2001) | L | |
| | $1.5\times10^{-3}$ | 3600 | Staudinger and Roberts (1996) | L | |
| | $1.7\times10^{-3}$ | | Mackay and Shiu (1981) | L | |
| | $2.0\times10^{-3}$ | 3900 | Hiatt (2013) | M | |
| | $1.9\times10^{-3}$ | 3300 | Chen et al. (2012) | M | |
| | $2.0\times10^{-3}$ | | Bobadilla et al. (2003) | M | |
| | $1.6\times10^{-3}$ | 3900 | Görgényi et al. (2002) | M | 664 |
| | $2.2\times10^{-3}$ | | Hovorka and Dohnal (1997) | M | 12 |
| | $1.8\times10^{-3}$ | 2600 | Kondoh and Nakajima (1997) | M | |
| | $2.0\times10^{-3}$ | 4300 | Dewulf et al. (1995) | M | |
| | $1.5\times10^{-3}$ | 4900 | Wright et al. (1992) | M | 665 |
| | $1.7\times10^{-3}$ | 3700 | Tse et al. (1992) | M | |
| | $1.7\times10^{-3}$ | 2100 | Lamarche and Droste (1989) | M | 345 |
| | $1.5\times10^{-3}$ | 3100 | Ashworth et al. (1988) | M | 278 |
| | $1.8\times10^{-3}$ | 4100 | Gossett (1987) | M | |
| | $1.3\times10^{-3}$ | 4900 | Ervin et al. (1980) | M | |
| | $1.8\times10^{-3}$ | | Warner et al. (1980) | M | |
| | $1.0\times10^{-3}$ | | Sato and Nakajima (1979b) | M | 14 |





Table A6.1: Chlorocarbons (C, H, Cl) (... continued)

| Substance Formula (Trivial Name) [CAS Registry Number] InChIKey | $H_s^{cp}$ (at $T^\ominus$) $\left[\dfrac{\text{mol}}{\text{m}^3\,\text{Pa}}\right]$ | $\dfrac{\text{d}\ln H_s^{cp}}{\text{d}(1/T)}$ [K] | Reference | Type | Note |
|---|---|---|---|---|---|
| | $1.8\times10^{-3}$ | 4400 | Rex (1906) | M | |
| | $1.7\times10^{-3}$ | | Mackay et al. (2006b) | V | |
| | $1.6\times10^{-3}$ | | Mackay et al. (1993) | V | |
| | $1.8\times10^{-3}$ | | Warner et al. (1980) | V | |
| | $1.7\times10^{-3}$ | | Smith and Bomberger (1980) | V | 24 |
| | $1.7\times10^{-3}$ | | Dilling (1977) | V | |
| | $1.7\times10^{-3}$ | | Hine and Mookerjee (1975) | V | |
| | $1.7\times10^{-3}$ | | Yaws (2003) | X | 237 |
| | $1.7\times10^{-3}$ | 3800 | Barr and Newsham (1987) | X | 298 |
| | $1.8\times10^{-3}$ | 1700 | Goldstein (1982) | X | 298 |
| | $2.4\times10^{-3}$ | | Ryan et al. (1988) | C | |
| | $1.8\times10^{-3}$ | | Shen (1982) | C | |
| | $5.1\times10^{-4}$ | | Wang et al. (2017) | Q | 80, 238 |
| | $4.2\times10^{-3}$ | | Wang et al. (2017) | Q | 80, 239 |
| | $6.5\times10^{-3}$ | | Wang et al. (2017) | Q | 80, 240 |
| | $2.6\times10^{-3}$ | | Gharagheizi et al. (2012) | Q | |
| | $2.0\times10^{-3}$ | | Raventos-Duran et al. (2010) | Q | 242, 243 |
| | $3.1\times10^{-3}$ | | Raventos-Duran et al. (2010) | Q | 244 |
| | $7.8\times10^{-4}$ | | Raventos-Duran et al. (2010) | Q | 245 |
| | $1.6\times10^{-3}$ | | Gharagheizi et al. (2010) | Q | 246 |
| | $3.2\times10^{-3}$ | | Hilal et al. (2008) | Q | |
| | $1.4\times10^{-3}$ | | Modarresi et al. (2007) | Q | 67 |
| | | 3300 | Kühne et al. (2005) | Q | |
| | $1.8\times10^{-3}$ | | Yaffe et al. (2003) | Q | 248, 249 |
| | $5.0\times10^{-4}$ | | English and Carroll (2001) | Q | 230, 274 |
| | $1.1\times10^{-3}$ | | Katritzky et al. (1998) | Q | |
| | $1.4\times10^{-3}$ | | Nirmalakhandan and Speece (1988) | Q | |
| | $1.8\times10^{-3}$ | | Mackay et al. (2006b) | ? | |
| | | 3900 | Kühne et al. (2005) | ? | |
| | $1.7\times10^{-3}$ | | Yaws (1999) | ? | 21 |
| | $1.1\times10^{-3}$ | | Abraham and Weathersby (1994) | ? | 21 |
| | $1.6\times10^{-3}$ | | Mackay et al. (1993) | ? | |
| | $1.7\times10^{-3}$ | | Yaws and Yang (1992) | ? | 21 |
| | $1.7\times10^{-3}$ | | Abraham et al. (1990) | ? | |
| 1,2-dichloroethane $CH_2ClCH_2Cl$ [107-06-2] WSLDOOZREJYCGB-UHFFFAOYSA-N | $7.5\times10^{-3}$ | 4400 | Schwardt et al. (2021) | L | 1 |
| | $8.9\times10^{-3}$ | 4300 | Burkholder et al. (2019) | L | |
| | $7.6\times10^{-3}$ | 3700 | Burkholder et al. (2019) | L | 70 |
| | $8.9\times10^{-3}$ | 4300 | Burkholder et al. (2015) | L | |
| | $7.6\times10^{-3}$ | 3700 | Burkholder et al. (2015) | L | 70 |
| | $8.4\times10^{-3}$ | 4200 | Brockbank (2013) | L | 1 |
| | $8.9\times10^{-3}$ | 4300 | Warneck (2007) | L | |
| | $9.1\times10^{-3}$ | 4300 | Fogg and Sangster (2003) | L | |
| | $7.8\times10^{-3}$ | 4200 | Staudinger and Roberts (2001) | L | |
| | $7.1\times10^{-3}$ | 4200 | Staudinger and Roberts (1996) | L | |
| | $9.1\times10^{-3}$ | | Mackay and Shiu (1981) | L | |
| | $8.2\times10^{-3}$ | 4400 | Hiatt (2013) | M | |
| | $9.1\times10^{-3}$ | 6100 | Chen et al. (2012) | M | |





Table A6.1: Chlorocarbons (C, H, Cl) (... continued)

| Substance Formula (Trivial Name) [CAS Registry Number] InChIKey | $H_s^{cp}$ (at $T^\ominus$) $\left[\dfrac{\mathrm{mol}}{\mathrm{m^3\,Pa}}\right]$ | $\dfrac{\mathrm{d}\ln H_s^{cp}}{\mathrm{d}(1/T)}$ [K] | Reference | Type | Note |
|---|---|---|---|---|---|
| | $5.4\times10^{-3}$ | | Ayuttaya et al. (2001) | M | 340 |
| | $5.7\times10^{-4}$ | | Ayuttaya et al. (2001) | M | 341 |
| | $4.2\times10^{-3}$ | | Ayuttaya et al. (2001) | M | 342 |
| | $8.1\times10^{-3}$ | | Ayuttaya et al. (2001) | M | 343 |
| | $7.5\times10^{-3}$ | | Welke et al. (1998) | M | |
| | $1.1\times10^{-2}$ | | Hovorka and Dohnal (1997) | M | 12 |
| | $6.2\times10^{-3}$ | 3700 | Kondoh and Nakajima (1997) | M | |
| | $9.3\times10^{-3}$ | 4600 | Dewulf et al. (1995) | M | |
| | $8.3\times10^{-3}$ | | Hoff et al. (1993) | M | |
| | $8.2\times10^{-3}$ | | Li et al. (1993) | M | |
| | $8.4\times10^{-3}$ | 4300 | Wright et al. (1992) | M | 666 |
| | $8.0\times10^{-3}$ | 3600 | Tse et al. (1992) | M | |
| | $6.4\times10^{-3}$ | 4500 | Bissonette et al. (1990) | M | |
| | $5.8\times10^{-3}$ | 3000 | Lamarche and Droste (1989) | M | 345 |
| | $7.6\times10^{-3}$ | | Guitart et al. (1989) | M | 14 |
| | $6.4\times10^{-3}$ | 1500 | Ashworth et al. (1988) | M | 33, 278 |
| | $8.4\times10^{-3}$ | 3500 | Leighton and Calo (1981) | M | |
| | $9.0\times10^{-3}$ | | Warner et al. (1980) | M | |
| | $4.4\times10^{-3}$ | | Sato and Nakajima (1979b) | M | 14 |
| | $1.1\times10^{-2}$ | | Pearson and McConnell (1975) | M | 649, 12 |
| | $7.9\times10^{-3}$ | 4400 | Hartkopf and Karger (1973) | M | |
| | $7.2\times10^{-3}$ | | Saylor et al. (1938) | M | 38 |
| | $8.6\times10^{-3}$ | 4400 | Rex (1906) | M | |
| | $8.2\times10^{-3}$ | | Mackay et al. (2006b) | V | |
| | $8.3\times10^{-3}$ | | Mackay et al. (1993) | V | |
| | $7.3\times10^{-3}$ | | Warner et al. (1980) | V | |
| | $8.1\times10^{-3}$ | | Dilling (1977) | V | |
| | $7.5\times10^{-3}$ | | Hine and Mookerjee (1975) | V | |
| | $8.3\times10^{-3}$ | | Yaws (2003) | X | 237 |
| | $8.5\times10^{-3}$ | 3700 | Barr and Newsham (1987) | X | 298 |
| | $9.0\times10^{-3}$ | 2400 | Goldstein (1982) | X | 298 |
| | $8.6\times10^{-3}$ | | Harrison et al. (1993) | C | |
| | $9.0\times10^{-3}$ | | Harrison et al. (1993) | C | |
| | $1.1\times10^{-2}$ | | Ryan et al. (1988) | C | |
| | $9.0\times10^{-3}$ | | Shen (1982) | C | |
| | $1.0\times10^{-2}$ | | Dilling (1977) | C | |
| | $1.1\times10^{-2}$ | | Hayer et al. (2022) | Q | 20 |
| | $1.2\times10^{-2}$ | | Keshavarz et al. (2022) | Q | |
| | $3.8\times10^{-3}$ | | Duchowicz et al. (2020) | Q | 184 |
| | $1.5\times10^{-3}$ | | Wang et al. (2017) | Q | 80, 238 |
| | $1.2\times10^{-2}$ | | Wang et al. (2017) | Q | 80, 239 |
| | $2.0\times10^{-2}$ | | Wang et al. (2017) | Q | 80, 240 |
| | $7.5\times10^{-3}$ | | Li et al. (2014) | Q | 241 |
| | $6.6\times10^{-3}$ | | Gharagheizi et al. (2012) | Q | |
| | $4.9\times10^{-3}$ | | Raventos-Duran et al. (2010) | Q | 271, 243 |
| | $9.9\times10^{-3}$ | | Raventos-Duran et al. (2010) | Q | 244 |
| | $7.8\times10^{-4}$ | | Raventos-Duran et al. (2010) | Q | 245 |



Table A6.1: Chlorocarbons (C, H, Cl) (. . . continued)

| Substance<br>Formula<br>(Trivial Name)<br>[CAS Registry Number]<br>InChIKey | $H_s^{cp}$<br>(at $T^{\ominus}$)<br>$\left[\dfrac{\mathrm{mol}}{\mathrm{m^3\,Pa}}\right]$ | $\dfrac{\mathrm{d}\ln H_s^{cp}}{\mathrm{d}(1/T)}$<br><br>[K] | Reference | Type | Note |
|---|---|---|---|---|---|
| | $1.1\times10^{-2}$ | | Gharagheizi et al. (2010) | Q | 246 |
| | $1.0\times10^{-2}$ | | Hilal et al. (2008) | Q | |
| | $5.1\times10^{-3}$ | | Modarresi et al. (2007) | Q | 67 |
| | | 3300 | Kühne et al. (2005) | Q | |
| | $8.2\times10^{-3}$ | | Yaffe et al. (2003) | Q | 248, 249 |
| | $3.4\times10^{-3}$ | | English and Carroll (2001) | Q | 230, 231 |
| | $2.2\times10^{-3}$ | | Katritzky et al. (1998) | Q | |
| | $7.7\times10^{-3}$ | | Russell et al. (1992) | Q | 279 |
| | $1.8\times10^{-3}$ | | Nirmalakhandan and Speece (1988) | Q | |
| | $8.4\times10^{-3}$ | | Duchowicz et al. (2020) | ? | 185, 21 |
| | $4.2\times10^{-3}$ | | MacBean (2012a) | ? | |
| | $7.0\times10^{-3}$ | | Mackay et al. (2006b) | ? | |
| | | 3600 | Kühne et al. (2005) | ? | |
| | $8.4\times10^{-3}$ | | Yaws (1999) | ? | 21 |
| | $4.5\times10^{-3}$ | | Abraham and Weathersby (1994) | ? | 21 |
| | $7.0\times10^{-3}$ | | Mackay et al. (1993) | ? | |
| | $8.3\times10^{-3}$ | | Yaws and Yang (1992) | ? | 21 |
| | $8.2\times10^{-3}$ | | Abraham et al. (1990) | ? | |
| | $1.2\times10^{-2}$ | | Chiou et al. (1980) | ? | 79 |
| 1,2-dichloroethane-d4<br>$CD_2ClCD_2Cl$<br>[17060-07-0]<br>WSLDOOZREJYCGB-LNLMKGTHSA-N | $8.7\times10^{-3}$ | 4300 | Hiatt (2013) | M | |
| 1,1,1-trichloroethane<br>$CH_3CCl_3$<br>(methylchloroform; MCF)<br>[71-55-6]<br>UOCLXMDMGBRAIB-UHFFFAOYSA-N | $6.0\times10^{-4}$ | 3800 | Schwardt et al. (2021) | L | 1 |
| | $6.0\times10^{-4}$ | 3700 | Burkholder et al. (2019) | L | |
| | $4.2\times10^{-4}$ | 4200 | Burkholder et al. (2019) | L | 70 |
| | $6.0\times10^{-4}$ | 3700 | Burkholder et al. (2015) | L | |
| | $4.2\times10^{-4}$ | 4200 | Burkholder et al. (2015) | L | 70 |
| | $5.8\times10^{-4}$ | 4100 | Brockbank (2013) | L | 1 |
| | $6.0\times10^{-4}$ | 3700 | Warneck (2007) | L | |
| | $6.2\times10^{-4}$ | 3900 | Fogg and Sangster (2003) | L | |
| | $5.9\times10^{-4}$ | 4000 | Staudinger and Roberts (2001) | L | |
| | $5.8\times10^{-4}$ | 3900 | Staudinger and Roberts (1996) | L | |
| | $3.6\times10^{-4}$ | | Mackay and Shiu (1981) | L | |
| | $6.9\times10^{-4}$ | 4000 | Hiatt (2013) | M | |
| | $5.4\times10^{-4}$ | 4100 | Chen et al. (2012) | M | |
| | $6.2\times10^{-4}$ | 3500 | Vane and Giroux (2000) | M | |
| | $7.1\times10^{-4}$ | | Chiang et al. (1998) | M | 12 |
| | $7.9\times10^{-4}$ | | Hovorka and Dohnal (1997) | M | 12 |
| | $6.7\times10^{-4}$ | 1900 | Kondoh and Nakajima (1997) | M | |
| | $4.8\times10^{-4}$ | | Turner et al. (1996) | M | |
| | $6.7\times10^{-4}$ | 4100 | Dewulf et al. (1995) | M | |
| | $5.5\times10^{-4}$ | 2500 | Robbins et al. (1993) | M | 667 |
| | $5.3\times10^{-4}$ | | Hoff et al. (1993) | M | |
| | $5.9\times10^{-4}$ | 3100 | Hansen et al. (1993) | M | 281 |
| | $5.7\times10^{-4}$ | | Li et al. (1993) | M | |



Table A6.1: Chlorocarbons (C, H, Cl) (...continued)

| Substance / Formula / (Trivial Name) / [CAS Registry Number] / InChIKey | $H_s^{cp}$ (at $T^\ominus$) $\left[\dfrac{\mathrm{mol}}{\mathrm{m^3\,Pa}}\right]$ | $\dfrac{\mathrm{d}\ln H_s^{cp}}{\mathrm{d}(1/T)}$ [K] | Reference | Type | Note |
|---|---|---|---|---|---|
| | $6.0\times10^{-4}$ | 3300 | Wright et al. (1992) | M | 668 |
| | $6.3\times10^{-4}$ | 3700 | Tse et al. (1992) | M | |
| | $7.9\times10^{-4}$ | 1300 | Kolb et al. (1992) | M | 277 |
| | $5.1\times10^{-4}$ | 5200 | Bissonette et al. (1990) | M | |
| | $3.2\times10^{-4}$ | | Guitart et al. (1989) | M | 14 |
| | $5.7\times10^{-4}$ | 3400 | Ashworth et al. (1988) | M | 278 |
| | $5.9\times10^{-4}$ | 4100 | Gossett (1987) | M | |
| | $5.8\times10^{-4}$ | 4100 | Munz and Roberts (1987) | M | |
| | $6.3\times10^{-4}$ | | Yurteri et al. (1987) | M | 12 |
| | $5.7\times10^{-4}$ | 4200 | Gossett et al. (1985) | M | |
| | $5.9\times10^{-4}$ | 4300 | Lincoff and Gossett (1984) | M | |
| | $7.6\times10^{-4}$ | 3200 | Hunter-Smith et al. (1983) | M | 658 |
| | $4.9\times10^{-4}$ | 4400 | Leighton and Calo (1981) | M | |
| | $2.7\times10^{-4}$ | 7000 | Ervin et al. (1980) | M | |
| | $2.0\times10^{-3}$ | | Warner et al. (1980) | M | |
| | $3.6\times10^{-4}$ | | Sato and Nakajima (1979b) | M | 14 |
| | $2.9\times10^{-4}$ | | Pearson and McConnell (1975) | M | 649, 12 |
| | $5.9\times10^{-4}$ | | Mackay et al. (2006b) | V | |
| | $6.8\times10^{-4}$ | | Mackay et al. (1993) | V | |
| | $7.0\times10^{-4}$ | 4700 | McLinden (1989) | V | |
| | $2.4\times10^{-3}$ | | Warner et al. (1980) | V | |
| | $3.4\times10^{-4}$ | | Dilling (1977) | V | 651 |
| | $4.0\times10^{-4}$ | | Dilling (1977) | V | 12 |
| | $1.1\times10^{-3}$ | | Dilling (1977) | V | 153 |
| | $6.1\times10^{-4}$ | | Hine and Mookerjee (1975) | V | |
| | $5.9\times10^{-4}$ | | Dilling et al. (1975) | V | |
| | $4.5\times10^{-4}$ | | Yaws (2003) | X | 237 |
| | $5.8\times10^{-4}$ | 4000 | Barr and Newsham (1987) | X | 298 |
| | $2.2\times10^{-3}$ | 1700 | Goldstein (1982) | X | 298 |
| | $3.1\times10^{-4}$ | | Ryan et al. (1988) | C | |
| | $2.0\times10^{-3}$ | | Shen (1982) | C | |
| | $2.3\times10^{-4}$ | | Wang et al. (2017) | Q | 80, 238 |
| | $7.6\times10^{-4}$ | | Wang et al. (2017) | Q | 80, 239 |
| | $2.0\times10^{-3}$ | | Wang et al. (2017) | Q | 80, 240 |
| | $2.6\times10^{-3}$ | | Gharagheizi et al. (2012) | Q | |
| | $1.6\times10^{-3}$ | | Raventos-Duran et al. (2010) | Q | 242, 243 |
| | $7.8\times10^{-4}$ | | Raventos-Duran et al. (2010) | Q | 244 |
| | $2.5\times10^{-3}$ | | Raventos-Duran et al. (2010) | Q | 245 |
| | $4.9\times10^{-4}$ | | Gharagheizi et al. (2010) | Q | 246 |
| | $9.0\times10^{-4}$ | | Hilal et al. (2008) | Q | |
| | $1.3\times10^{-3}$ | | Modarresi et al. (2007) | Q | 67 |
| | | 3700 | Kühne et al. (2005) | Q | |
| | $6.1\times10^{-4}$ | | Yaffe et al. (2003) | Q | 248, 249 |
| | $7.9\times10^{-4}$ | | English and Carroll (2001) | Q | 230, 260 |
| | $5.4\times10^{-4}$ | | Katritzky et al. (1998) | Q | |
| | $2.3\times10^{-3}$ | | Nirmalakhandan and Speece (1988) | Q | |
| | $1.6\times10^{-3}$ | | Arbuckle (1983) | Q | |



Table A6.1: Chlorocarbons (C, H, Cl) (...continued)

| Substance Formula (Trivial Name) [CAS Registry Number] InChIKey | $H_s^{cp}$ (at $T^\ominus$) $\left[\dfrac{\text{mol}}{\text{m}^3\,\text{Pa}}\right]$ | $\dfrac{\text{d}\ln H_s^{cp}}{\text{d}(1/T)}$ [K] | Reference | Type | Note |
|---|---|---|---|---|---|
| | $5.7\times10^{-4}$ | | Mackay et al. (2006b) | ? | |
| | | 3700 | Kühne et al. (2005) | ? | |
| | $4.6\times10^{-4}$ | | Yaws (1999) | ? | 21 |
| | $3.8\times10^{-4}$ | | Abraham and Weathersby (1994) | ? | 21 |
| | $5.7\times10^{-4}$ | | Mackay et al. (1993) | ? | |
| | $5.6\times10^{-4}$ | | Abraham et al. (1990) | ? | |
| | $1.6\times10^{-3}$ | | Chiou et al. (1980) | ? | 79 |
| 1,1,2-trichloroethane | $1.1\times10^{-2}$ | 4400 | Schwardt et al. (2021) | L | 1 |
| $CHCl_2CH_2Cl$ | $1.1\times10^{-2}$ | 4100 | Burkholder et al. (2019) | L | |
| [79-00-5] | $1.1\times10^{-2}$ | 4100 | Burkholder et al. (2015) | L | |
| UBOXGVDOUJQMTN-UHFFFAOYSA-N | $1.1\times10^{-2}$ | 4400 | Brockbank (2013) | L | 1, 669 |
| | $1.1\times10^{-2}$ | 4100 | Warneck (2007) | L | |
| | $1.2\times10^{-2}$ | 4200 | Fogg and Sangster (2003) | L | |
| | $1.1\times10^{-2}$ | 4900 | Staudinger and Roberts (2001) | L | |
| | $1.1\times10^{-2}$ | 4900 | Staudinger and Roberts (1996) | L | |
| | $8.3\times10^{-3}$ | | Mackay and Shiu (1981) | L | |
| | $7.3\times10^{-3}$ | 2400 | Schwardt et al. (2021) | M | 670, 11 |
| | $1.4\times10^{-2}$ | 5400 | Hiatt (2013) | M | |
| | $1.2\times10^{-2}$ | | Bobadilla et al. (2003) | M | |
| | $1.1\times10^{-2}$ | 4700 | Dewulf et al. (1999) | M | |
| | $1.5\times10^{-2}$ | | Dohnal and Hovorka (1999) | M | 12 |
| | $1.5\times10^{-2}$ | | Hovorka and Dohnal (1997) | M | 12 |
| | $1.1\times10^{-2}$ | 5100 | Kondoh and Nakajima (1997) | M | |
| | $1.2\times10^{-2}$ | 5900 | Hansen et al. (1993) | M | 281 |
| | $1.1\times10^{-2}$ | 4300 | Wright et al. (1992) | M | 671 |
| | $1.1\times10^{-2}$ | 4100 | Tse et al. (1992) | M | |
| | $1.0\times10^{-2}$ | 4800 | Ashworth et al. (1988) | M | 278 |
| | $1.2\times10^{-2}$ | 3700 | Leighton and Calo (1981) | M | |
| | $6.6\times10^{-3}$ | | Sato and Nakajima (1979b) | M | 14 |
| | $1.1\times10^{-2}$ | | Mackay et al. (2006b) | V | |
| | $1.0\times10^{-2}$ | | Mackay et al. (1993) | V | |
| | $1.1\times10^{-2}$ | | Dilling (1977) | V | |
| | $1.1\times10^{-2}$ | | Hine and Mookerjee (1975) | V | |
| | $1.1\times10^{-2}$ | | Yaws (2003) | X | 237 |
| | $1.1\times10^{-2}$ | 4300 | Barr and Newsham (1987) | X | 298 |
| | $1.2\times10^{-2}$ | 2700 | Goldstein (1982) | X | 298 |
| | $1.3\times10^{-3}$ | | Ryan et al. (1988) | C | |
| | $1.2\times10^{-2}$ | | Keshavarz et al. (2022) | Q | |
| | $7.7\times10^{-3}$ | | Duchowicz et al. (2020) | Q | |
| | $4.9\times10^{-3}$ | | Wang et al. (2017) | Q | 80, 238 |
| | $2.1\times10^{-2}$ | | Wang et al. (2017) | Q | 80, 239 |
| | $4.1\times10^{-2}$ | | Wang et al. (2017) | Q | 80, 240 |
| | $1.1\times10^{-2}$ | | Li et al. (2014) | Q | 241 |
| | $1.2\times10^{-2}$ | | Raventos-Duran et al. (2010) | Q | 271, 243 |
| | $1.6\times10^{-2}$ | | Raventos-Duran et al. (2010) | Q | 244 |
| | $2.5\times10^{-3}$ | | Raventos-Duran et al. (2010) | Q | 245 |
| | $1.0\times10^{-2}$ | | Gharagheizi et al. (2010) | Q | 246 |



Table A6.1: Chlorocarbons (C, H, Cl) (. . . continued)

| Substance<br>Formula<br>(Trivial Name)<br>[CAS Registry Number]<br>InChIKey | $H_s^{cp}$<br>(at $T^{\ominus}$)<br>$\left[\dfrac{\text{mol}}{\text{m}^3\,\text{Pa}}\right]$ | $\dfrac{\text{d}\ln H_s^{cp}}{\text{d}(1/T)}$<br><br>[K] | Reference | Type | Note |
|---|---|---|---|---|---|
| | $1.5\times10^{-2}$ | | Hilal et al. (2008) | Q | |
| | $6.7\times10^{-3}$ | | Modarresi et al. (2007) | Q | 67 |
| | | 3700 | Kühne et al. (2005) | Q | |
| | $8.6\times10^{-3}$ | | Yaffe et al. (2003) | Q | 248, 249 |
| | $1.9\times10^{-3}$ | | Katritzky et al. (1998) | Q | |
| | $3.3\times10^{-3}$ | | Nirmalakhandan and Speece (1988) | Q | |
| | $7.6\times10^{-3}$ | | Arbuckle (1983) | Q | |
| | $1.2\times10^{-2}$ | | Duchowicz et al. (2020) | ? | 185, 21 |
| | $1.1\times10^{-2}$ | | Mackay et al. (2006b) | ? | |
| | | 4200 | Kühne et al. (2005) | ? | |
| | $1.1\times10^{-2}$ | | Yaws (1999) | ? | 21 |
| | $6.9\times10^{-3}$ | | Abraham and Weathersby (1994) | ? | 21 |
| | $1.1\times10^{-2}$ | | Mackay et al. (1993) | ? | |
| | $1.0\times10^{-2}$ | | Yaws and Yang (1992) | ? | 21 |
| | $1.2\times10^{-2}$ | | Abraham et al. (1990) | ? | |
| 1,1,2-trichloroethane-d3<br>CDCl$_2$CD$_2$Cl<br>[171086-93-4]<br>UBOXGVDOUJQMTN-FUDHJZNOSA-N | $1.3\times10^{-2}$ | 5100 | Hiatt (2013) | M | |
| 1,1,1,2-tetrachloroethane<br>CCl$_3$CH$_2$Cl<br>[630-20-6]<br>QVLAWKAXOMEXPM-UHFFFAOYSA-N | $5.0\times10^{-3}$ | 6700 | Schwardt et al. (2021) | L | 1 |
| | $4.2\times10^{-3}$ | 4600 | Burkholder et al. (2019) | L | |
| | $4.2\times10^{-3}$ | 4600 | Burkholder et al. (2015) | L | |
| | $4.2\times10^{-3}$ | 4900 | Brockbank (2013) | L | 1 |
| | $4.2\times10^{-3}$ | 4600 | Warneck (2007) | L | |
| | $2.4\times10^{-2}$ | 3200 | Staudinger and Roberts (2001) | L | |
| | $3.6\times10^{-3}$ | | Mackay and Shiu (1981) | L | |
| | $4.5\times10^{-3}$ | 11000 | Schwardt et al. (2021) | M | 672 |
| | $4.8\times10^{-3}$ | 4800 | Hiatt (2013) | M | |
| | $4.3\times10^{-3}$ | 4100 | Kondoh and Nakajima (1997) | M | |
| | $4.0\times10^{-3}$ | 4400 | Wright et al. (1992) | M | 673 |
| | $4.5\times10^{-3}$ | 4600 | Tse et al. (1992) | M | |
| | $2.1\times10^{-3}$ | | Sato and Nakajima (1979b) | M | 14 |
| | $4.0\times10^{-3}$ | | Mackay et al. (2006b) | V | |
| | $4.2\times10^{-3}$ | 5000 | Fogg and Sangster (2003) | V | |
| | $4.1\times10^{-3}$ | | Mackay et al. (1993) | V | |
| | $3.7\times10^{-3}$ | | Dilling (1977) | V | |
| | $4.1\times10^{-3}$ | | Yaws (2003) | X | 237 |
| | $1.2\times10^{-2}$ | | Keshavarz et al. (2022) | Q | |
| | $3.1\times10^{-3}$ | | Duchowicz et al. (2020) | Q | 184 |
| | $1.1\times10^{-2}$ | | Gharagheizi et al. (2012) | Q | |
| | $7.8\times10^{-3}$ | | Raventos-Duran et al. (2010) | Q | 271, 243 |
| | $3.9\times10^{-3}$ | | Raventos-Duran et al. (2010) | Q | 244 |
| | $6.2\times10^{-3}$ | | Raventos-Duran et al. (2010) | Q | 245 |
| | $3.9\times10^{-3}$ | | Gharagheizi et al. (2010) | Q | 246 |
| | $3.9\times10^{-3}$ | | Hilal et al. (2008) | Q | |
| | $6.0\times10^{-3}$ | | Modarresi et al. (2007) | Q | 67 |



Table A6.1: Chlorocarbons (C, H, Cl) (. . . continued)

| Substance Formula (Trivial Name) [CAS Registry Number] InChIKey | $H_s^{cp}$ (at $T^\ominus$) $\left[\dfrac{\text{mol}}{\text{m}^3\,\text{Pa}}\right]$ | $\dfrac{\text{d}\ln H_s^{cp}}{\text{d}(1/T)}$ [K] | Reference | Type | Note |
|---|---|---|---|---|---|
| | | 4100 | Kühne et al. (2005) | Q | |
| | $3.8\times10^{-3}$ | | Yaffe et al. (2003) | Q | 248, 249 |
| | $4.5\times10^{-3}$ | | English and Carroll (2001) | Q | 230, 231 |
| | $1.2\times10^{-3}$ | | Katritzky et al. (1998) | Q | |
| | $5.4\times10^{-3}$ | | Nirmalakhandan et al. (1997) | Q | |
| | $3.9\times10^{-3}$ | | Duchowicz et al. (2020) | ? | 185, 21 |
| | | 4600 | Kühne et al. (2005) | ? | |
| | $4.1\times10^{-3}$ | | Yaws (1999) | ? | 21 |
| | $2.2\times10^{-3}$ | | Abraham and Weathersby (1994) | ? | 21 |
| | $3.5\times10^{-3}$ | | Abraham et al. (1990) | ? | |
| 1,1,2,2-tetrachloroethane $CHCl_2CHCl_2$ [79-34-5] QPFMBZIOSGYJDE-UHFFFAOYSA-N | $2.3\times10^{-2}$ | 3700 | Schwardt et al. (2021) | L | 1 |
| | $2.3\times10^{-2}$ | 4800 | Burkholder et al. (2019) | L | |
| | $2.3\times10^{-2}$ | 4800 | Burkholder et al. (2015) | L | |
| | $2.4\times10^{-2}$ | 5000 | Brockbank (2013) | L | 1 |
| | $2.4\times10^{-2}$ | 4800 | Warneck (2007) | L | |
| | $2.4\times10^{-2}$ | 3200 | Staudinger and Roberts (1996) | L | |
| | $2.1\times10^{-2}$ | | Mackay and Shiu (1981) | L | |
| | $3.3\times10^{-2}$ | 7200 | Hiatt (2013) | M | |
| | $3.0\times10^{-2}$ | | Hovorka and Dohnal (1997) | M | 12 |
| | $2.3\times10^{-2}$ | 6800 | Kondoh and Nakajima (1997) | M | 674 |
| | $2.9\times10^{-2}$ | | Li and Carr (1993) | M | |
| | $2.2\times10^{-2}$ | 3100 | Wright et al. (1992) | M | 675 |
| | $2.6\times10^{-2}$ | 4800 | Tse et al. (1992) | M | |
| | $2.2\times10^{-2}$ | 2900 | Ashworth et al. (1988) | M | 42, 278 |
| | $2.7\times10^{-2}$ | 3500 | Leighton and Calo (1981) | M | |
| | $1.4\times10^{-2}$ | | Sato and Nakajima (1979b) | M | 14 |
| | $2.1\times10^{-2}$ | | Mackay et al. (2006b) | V | |
| | $2.2\times10^{-2}$ | | Mackay et al. (1993) | V | |
| | $2.1\times10^{-2}$ | | Dilling (1977) | V | |
| | $2.2\times10^{-2}$ | | Hine and Mookerjee (1975) | V | |
| | $2.8\times10^{-2}$ | | Yaws (2003) | X | 237 |
| | $1.8\times10^{-2}$ | 4200 | Barr and Newsham (1987) | X | 298 |
| | $2.3\times10^{-2}$ | 3000 | Goldstein (1982) | X | 298 |
| | $2.5\times10^{-2}$ | | Ryan et al. (1988) | C | |
| | $1.2\times10^{-2}$ | | Keshavarz et al. (2022) | Q | |
| | $1.5\times10^{-2}$ | | Duchowicz et al. (2020) | Q | 299 |
| | $1.6\times10^{-2}$ | | Wang et al. (2017) | Q | 80, 238 |
| | $2.7\times10^{-2}$ | | Wang et al. (2017) | Q | 80, 239 |
| | $9.3\times10^{-2}$ | | Wang et al. (2017) | Q | 80, 240 |
| | $3.9\times10^{-2}$ | | Raventos-Duran et al. (2010) | Q | 271, 243 |
| | $2.0\times10^{-2}$ | | Raventos-Duran et al. (2010) | Q | 244 |
| | $6.2\times10^{-3}$ | | Raventos-Duran et al. (2010) | Q | 245 |
| | $2.8\times10^{-2}$ | | Gharagheizi et al. (2010) | Q | 246 |
| | $1.9\times10^{-2}$ | | Hilal et al. (2008) | Q | |
| | $5.5\times10^{-3}$ | | Modarresi et al. (2007) | Q | 67 |
| | | 4100 | Kühne et al. (2005) | Q | |
| | $2.7\times10^{-2}$ | | Yaffe et al. (2003) | Q | 248, 249 |



Table A6.1: Chlorocarbons (C, H, Cl) (. . . continued)

| Substance<br>Formula<br>(Trivial Name)<br>[CAS Registry Number]<br>InChIKey | $H_s^{cp}$<br>(at $T^{\ominus}$)<br>$\left[\dfrac{\text{mol}}{\text{m}^3\,\text{Pa}}\right]$ | $\dfrac{\text{d}\ln H_s^{cp}}{\text{d}(1/T)}$<br><br>[K] | Reference | Type | Note |
|---|---|---|---|---|---|
| | $1.1\times10^{-3}$ | | Katritzky et al. (1998) | Q | |
| | $6.1\times10^{-3}$ | | Nirmalakhandan and Speece (1988) | Q | |
| | $2.7\times10^{-2}$ | | Duchowicz et al. (2020) | ? | 185, 21 |
| | $3.9\times10^{-2}$ | | Mackay et al. (2006b) | ? | |
| | | 4500 | Kühne et al. (2005) | ? | |
| | $2.8\times10^{-2}$ | | Yaws (1999) | ? | 21 |
| | $1.4\times10^{-2}$ | | Abraham and Weathersby (1994) | ? | 21 |
| | $3.9\times10^{-2}$ | | Mackay et al. (1993) | ? | |
| | $3.0\times10^{-2}$ | | Yaws and Yang (1992) | ? | 21 |
| | $2.6\times10^{-2}$ | | Abraham et al. (1990) | ? | |
| | $3.0\times10^{-2}$ | | Chiou et al. (1980) | ? | 79 |
| pentachloroethane<br>CHCl$_2$CCl$_3$<br>[76-01-7]<br>BNIXVQGCZULYKV-UHFFFAOYSA-N | $4.1\times10^{-3}$ | 7700 | Burkholder et al. (2019) | L | 676 |
| | $4.2\times10^{-3}$ | 7700 | Brockbank (2013) | L | 1 |
| | $4.5\times10^{-3}$ | | Mackay and Shiu (1981) | L | |
| | $5.9\times10^{-3}$ | 5400 | Hiatt (2013) | M | |
| | $5.1\times10^{-3}$ | | Duchowicz et al. (2020) | V | 186 |
| | $5.2\times10^{-3}$ | | HSDB (2015) | V | |
| | $4.1\times10^{-3}$ | | Mackay et al. (2006b) | V | |
| | $4.0\times10^{-3}$ | | Mackay et al. (1993) | V | |
| | $5.3\times10^{-3}$ | | Meylan and Howard (1991) | V | |
| | $4.0\times10^{-3}$ | | Dilling (1977) | V | |
| | $4.0\times10^{-3}$ | | Hine and Mookerjee (1975) | V | |
| | $6.2\times10^{-3}$ | | Duchowicz et al. (2020) | Q | |
| | $4.0\times10^{-3}$ | | Li et al. (2014) | Q | 241 |
| | $2.0\times10^{-2}$ | | Gharagheizi et al. (2012) | Q | |
| | $2.0\times10^{-2}$ | | Raventos-Duran et al. (2010) | Q | 242, 243 |
| | $4.9\times10^{-3}$ | | Raventos-Duran et al. (2010) | Q | 244 |
| | $2.0\times10^{-2}$ | | Raventos-Duran et al. (2010) | Q | 245 |
| | $6.1\times10^{-3}$ | | Hilal et al. (2008) | Q | |
| | $3.6\times10^{-3}$ | | Modarresi et al. (2007) | Q | 67 |
| | $4.0\times10^{-4}$ | | Katritzky et al. (1998) | Q | |
| | $1.9\times10^{-2}$ | | Meylan and Howard (1991) | Q | |
| | $1.0\times10^{-2}$ | | Nirmalakhandan and Speece (1988) | Q | |
| | $5.1\times10^{-3}$ | | Yaws (1999) | ? | 21 |
| | $5.4\times10^{-3}$ | | Yaws and Yang (1992) | ? | 21 |
| | $4.2\times10^{-3}$ | | Abraham et al. (1990) | ? | |
| hexachloroethane<br>C$_2$Cl$_6$<br>[67-72-1]<br>VHHHONWQHHHLTI-UHFFFAOYSA-N | $2.7\times10^{-3}$ | 5700 | Burkholder et al. (2019) | L | |
| | $2.6\times10^{-3}$ | 5600 | Burkholder et al. (2015) | L | |
| | $2.8\times10^{-3}$ | 6100 | Brockbank (2013) | L | 1 |
| | $2.5\times10^{-3}$ | 5600 | Staudinger and Roberts (1996) | L | |
| | $1.2\times10^{-3}$ | 2600 | Ashworth et al. (1988) | M | 33, 278 |
| | $2.5\times10^{-3}$ | 5600 | Munz and Roberts (1987) | M | |
| | $3.4\times10^{-3}$ | | Munz and Roberts (1986) | M | |
| | $1.0\times10^{-3}$ | | Warner et al. (1980) | M | |
| | $4.2\times10^{-3}$ | | Mackay et al. (2006b) | V | |
| | $3.6\times10^{-3}$ | | Lide and Frederikse (1995) | V | |





Table A6.1: Chlorocarbons (C, H, Cl) (...continued)

| Substance Formula (Trivial Name) [CAS Registry Number] InChIKey | $H_s^{cp}$ (at $T^{\ominus}$) $\left[\dfrac{\text{mol}}{\text{m}^3\,\text{Pa}}\right]$ | $\dfrac{\text{d}\ln H_s^{cp}}{\text{d}(1/T)}$ [K] | Reference | Type | Note |
|---|---|---|---|---|---|
| | $1.5\times10^{-2}$ | | Hwang et al. (1992) | V | |
| | $2.2\times10^{-4}$ | | Ballschmiter and Wittlinger (1991) | V | |
| | $7.7\times10^{-4}$ | | Mackay and Shiu (1981) | V | |
| | $8.1\times10^{-3}$ | | Dilling (1977) | V | |
| | $4.3\times10^{-3}$ | | Hine and Mookerjee (1975) | V | |
| | $3.9\times10^{-4}$ | | Yaws (2003) | X | 237 |
| | $1.0\times10^{-3}$ | 2100 | Goldstein (1982) | X | 298 |
| | $9.8\times10^{-4}$ | | Ryan et al. (1988) | C | |
| | $1.0\times10^{-3}$ | | Shen (1982) | C | |
| | $7.3\times10^{-4}$ | | Keshavarz et al. (2022) | Q | |
| | $2.5\times10^{-3}$ | | Duchowicz et al. (2020) | Q | |
| | $2.4\times10^{-3}$ | | Zhang et al. (2010) | Q | 287, 288 |
| | $1.8\times10^{-3}$ | | Zhang et al. (2010) | Q | 287, 289 |
| | $1.9\times10^{-3}$ | | Zhang et al. (2010) | Q | 287, 290 |
| | $3.9\times10^{-3}$ | | Zhang et al. (2010) | Q | 287, 291 |
| | $3.7\times10^{-4}$ | | Gharagheizi et al. (2010) | Q | 246 |
| | $3.9\times10^{-3}$ | | Hilal et al. (2008) | Q | |
| | $9.3\times10^{-4}$ | | Modarresi et al. (2007) | Q | 67 |
| | $8.0\times10^{-4}$ | | Yaffe et al. (2003) | Q | 248, 249 |
| | $2.4\times10^{-4}$ | | Yao et al. (2002) | Q | 229 |
| | $1.0\times10^{-3}$ | | Nirmalakhandan and Speece (1988) | Q | |
| | $2.5\times10^{-3}$ | | Duchowicz et al. (2020) | ? | 185, 21 |
| | $1.2\times10^{-3}$ | | Mackay et al. (2006b) | ? | |
| | $3.9\times10^{-4}$ | | Yaws (1999) | ? | 21 |
| | $1.2\times10^{-3}$ | | Mackay et al. (1993) | ? | |
| | $4.4\times10^{-4}$ | | Yaws and Yang (1992) | ? | 21 |
| 1-chloropropane $C_3H_7Cl$ [540-54-5] SNMVRZFUUCLYTO-UHFFFAOYSA-N | $7.6\times10^{-4}$ | 4500 | Brockbank (2013) | L | 1 |
| | $6.9\times10^{-4}$ | | Li et al. (1993) | M | |
| | $4.3\times10^{-4}$ | | Sato and Nakajima (1979b) | M | 14 |
| | $7.7\times10^{-4}$ | 4400 | Rex (1906) | M | |
| | $7.5\times10^{-4}$ | | Duchowicz et al. (2020) | V | 186 |
| | $7.6\times10^{-4}$ | | HSDB (2015) | V | |
| | $6.9\times10^{-4}$ | | Mackay et al. (2006b) | V | |
| | $7.1\times10^{-4}$ | | Mackay et al. (1993) | V | |
| | $7.1\times10^{-4}$ | | Abraham (1984) | V | |
| | $7.3\times10^{-4}$ | | Hine and Mookerjee (1975) | V | |
| | $9.2\times10^{-4}$ | | Yaws (2003) | X | 237, 12 |
| | $1.3\times10^{-3}$ | | Duchowicz et al. (2020) | Q | |
| | $7.3\times10^{-4}$ | | Li et al. (2014) | Q | 241 |
| | $4.3\times10^{-4}$ | | Gharagheizi et al. (2012) | Q | |
| | $4.9\times10^{-4}$ | | Raventos-Duran et al. (2010) | Q | 271, 243 |
| | $7.8\times10^{-4}$ | | Raventos-Duran et al. (2010) | Q | 244 |
| | $6.2\times10^{-4}$ | | Raventos-Duran et al. (2010) | Q | 245 |
| | $6.8\times10^{-4}$ | | Gharagheizi et al. (2010) | Q | 246 |
| | $1.1\times10^{-3}$ | | Hilal et al. (2008) | Q | |
| | $1.0\times10^{-3}$ | | Modarresi et al. (2007) | Q | 67 |
| | | 3300 | Kühne et al. (2005) | Q | |



Table A6.1: Chlorocarbons (C, H, Cl) (...continued)

| Substance<br>Formula<br>(Trivial Name)<br>[CAS Registry Number]<br>InChIKey | $H_s^{cp}$<br>(at $T^{\ominus}$)<br>$\left[\dfrac{\mathrm{mol}}{\mathrm{m^3\,Pa}}\right]$ | $\dfrac{\mathrm{d}\ln H_s^{cp}}{\mathrm{d}(1/T)}$<br><br>[K] | Reference | Type | Note |
|---|---|---|---|---|---|
| | | | Yaffe et al. (2003) | Q | 356 |
| | $2.8\times10^{-4}$ | | Yao et al. (2002) | Q | 229 |
| | $7.3\times10^{-4}$ | | English and Carroll (2001) | Q | 230, 231 |
| | $1.2\times10^{-3}$ | | Katritzky et al. (1998) | Q | |
| | $4.7\times10^{-4}$ | | Suzuki et al. (1992) | Q | 232 |
| | $6.2\times10^{-4}$ | | Nirmalakhandan and Speece (1988) | Q | |
| | | 3500 | Kühne et al. (2005) | ? | |
| | $7.5\times10^{-4}$ | | Yaws (1999) | ? | 21, 12 |
| | $4.4\times10^{-4}$ | | Abraham and Weatherby (1994) | ? | 21 |
| | $9.1\times10^{-4}$ | | Yaws and Yang (1992) | ? | 21, 12 |
| | $7.0\times10^{-4}$ | | Abraham et al. (1990) | ? | |
| 2-chloropropane<br>$C_3H_7Cl$<br>[75-29-6]<br>ULYZAYCEDJDHCC-UHFFFAOYSA-N | $5.7\times10^{-4}$ | 4400 | Brockbank (2013) | L | 1 |
| | $5.4\times10^{-4}$ | | Li et al. (1993) | M | |
| | $5.6\times10^{-4}$ | 4300 | Rex (1906) | M | |
| | $5.6\times10^{-4}$ | | Duchowicz et al. (2020) | V | 186 |
| | $5.5\times10^{-4}$ | | HSDB (2015) | V | |
| | $5.6\times10^{-4}$ | | Mackay et al. (2006b) | V | |
| | $5.5\times10^{-4}$ | | Mackay et al. (1993) | V | |
| | $6.1\times10^{-4}$ | | Hine and Mookerjee (1975) | V | |
| | $6.8\times10^{-4}$ | | Yaws (2003) | X | 237, 12 |
| | $5.9\times10^{-4}$ | | Duchowicz et al. (2020) | Q | |
| | $9.8\times10^{-4}$ | | Gharagheizi et al. (2012) | Q | |
| | $4.9\times10^{-4}$ | | Raventos-Duran et al. (2010) | Q | 271, 243 |
| | $4.9\times10^{-4}$ | | Raventos-Duran et al. (2010) | Q | 244 |
| | $6.2\times10^{-4}$ | | Raventos-Duran et al. (2010) | Q | 245 |
| | $4.8\times10^{-4}$ | | Gharagheizi et al. (2010) | Q | 246 |
| | $6.0\times10^{-4}$ | | Hilal et al. (2008) | Q | |
| | $8.1\times10^{-4}$ | | Modarresi et al. (2007) | Q | 67 |
| | $6.1\times10^{-4}$ | | Yaffe et al. (2003) | Q | 248, 249 |
| | $6.2\times10^{-4}$ | | English and Carroll (2001) | Q | 230, 260 |
| | $1.5\times10^{-3}$ | | Katritzky et al. (1998) | Q | |
| | $9.5\times10^{-4}$ | | Russell et al. (1992) | Q | 279 |
| | $4.1\times10^{-4}$ | | Suzuki et al. (1992) | Q | 232 |
| | $5.1\times10^{-4}$ | | Nirmalakhandan and Speece (1988) | Q | |
| | $5.6\times10^{-4}$ | | Yaws (1999) | ? | 21, 12 |
| | $6.8\times10^{-4}$ | | Yaws and Yang (1992) | ? | 21, 12 |
| | $6.1\times10^{-4}$ | | Abraham et al. (1990) | ? | |
| 1,1-dichloropropane<br>$C_3H_6Cl_2$<br>[78-99-9]<br>WIHMGGWNMISDNJ-UHFFFAOYSA-N | $2.6\times10^{-3}$ | | Duchowicz et al. (2020) | V | 186 |
| | $3.3\times10^{-3}$ | | Duchowicz et al. (2020) | V | 186 |
| | $2.6\times10^{-3}$ | | HSDB (2015) | V | |
| | $2.7\times10^{-3}$ | | Duchowicz et al. (2020) | Q | |
| | $2.7\times10^{-3}$ | | Duchowicz et al. (2020) | Q | |



Table A6.1: Chlorocarbons (C, H, Cl) (...continued)

| Substance Formula (Trivial Name) [CAS Registry Number] InChIKey | $H_s^{cp}$ (at $T^\ominus$) $\left[\dfrac{\text{mol}}{\text{m}^3\,\text{Pa}}\right]$ | $\dfrac{\text{d}\ln H_s^{cp}}{\text{d}(1/T)}$ [K] | Reference | Type | Note |
|---|---|---|---|---|---|
| 1,2-dichloropropane | $3.7\times10^{-3}$ | 4100 | Schwardt et al. (2021) | L | 1 |
| $C_3H_6Cl_2$ | $3.4\times10^{-3}$ | 4300 | Burkholder et al. (2019) | L | |
| [78-87-5] | $3.4\times10^{-3}$ | 4300 | Burkholder et al. (2015) | L | |
| KNKRKFALVUDBJE-UHFFFAOYSA-N | $3.4\times10^{-3}$ | 4000 | Brockbank (2013) | L | 1 |
| | $3.4\times10^{-3}$ | 4300 | Staudinger and Roberts (2001) | L | |
| | $3.4\times10^{-3}$ | 4300 | Staudinger and Roberts (1996) | L | |
| | $4.3\times10^{-3}$ | 4400 | Hiatt (2013) | M | |
| | $4.2\times10^{-3}$ | | Bobadilla et al. (2003) | M | |
| | $3.5\times10^{-3}$ | 4300 | Dewulf et al. (1999) | M | |
| | $4.4\times10^{-3}$ | | Dohnal and Hovorka (1999) | M | 12 |
| | $5.5\times10^{-3}$ | | Welke et al. (1998) | M | |
| | $4.6\times10^{-3}$ | | Hovorka and Dohnal (1997) | M | 12 |
| | $4.3\times10^{-3}$ | 3700 | Kondoh and Nakajima (1997) | M | |
| | $3.6\times10^{-3}$ | 4200 | Wright et al. (1992) | M | 677 |
| | $3.8\times10^{-3}$ | 3800 | Tse et al. (1992) | M | |
| | $3.0\times10^{-3}$ | 3800 | Bissonette et al. (1990) | M | |
| | $3.8\times10^{-3}$ | 4700 | Ashworth et al. (1988) | M | 33, 278 |
| | $4.9\times10^{-3}$ | | Albanese et al. (1987) | M | |
| | $3.4\times10^{-3}$ | 4300 | Leighton and Calo (1981) | M | |
| | $3.5\times10^{-3}$ | | Warner et al. (1980) | M | |
| | $2.1\times10^{-3}$ | | Sato and Nakajima (1979b) | M | 14 |
| | $3.7\times10^{-3}$ | | Mackay et al. (2006b) | V | |
| | $3.7\times10^{-3}$ | | Mackay et al. (1993) | V | |
| | $3.6\times10^{-3}$ | | Warner et al. (1980) | V | |
| | $3.4\times10^{-3}$ | | Hine and Mookerjee (1975) | V | |
| | $3.5\times10^{-3}$ | | Yaws (2003) | X | 237 |
| | $3.4\times10^{-3}$ | 2100 | Goldstein (1982) | X | 298 |
| | $3.5\times10^{-3}$ | | Horvath and Getzen (1999) | C | |
| | $3.4\times10^{-3}$ | | Ryan et al. (1988) | C | |
| | $3.5\times10^{-3}$ | | Shen (1982) | C | |
| | $1.6\times10^{-2}$ | | Keshavarz et al. (2022) | Q | |
| | $1.8\times10^{-3}$ | | Duchowicz et al. (2020) | Q | 184 |
| | $1.4\times10^{-3}$ | | Wang et al. (2017) | Q | 80, 238 |
| | $6.2\times10^{-3}$ | | Wang et al. (2017) | Q | 80, 239 |
| | $8.9\times10^{-3}$ | | Wang et al. (2017) | Q | 80, 240 |
| | $4.7\times10^{-3}$ | | Gharagheizi et al. (2012) | Q | |
| | $3.9\times10^{-3}$ | | Raventos-Duran et al. (2010) | Q | 242, 243 |
| | $4.9\times10^{-3}$ | | Raventos-Duran et al. (2010) | Q | 244 |
| | $6.2\times10^{-4}$ | | Raventos-Duran et al. (2010) | Q | 245 |
| | $4.1\times10^{-3}$ | | Gharagheizi et al. (2010) | Q | 246 |
| | $5.4\times10^{-3}$ | | Hilal et al. (2008) | Q | |
| | $3.6\times10^{-3}$ | | Modarresi et al. (2007) | Q | 67 |
| | | 3700 | Kühne et al. (2005) | Q | |
| | $3.8\times10^{-3}$ | | Yaffe et al. (2003) | Q | 248, 249 |
| | $2.1\times10^{-3}$ | | English and Carroll (2001) | Q | 230, 231 |
| | $2.8\times10^{-3}$ | | Katritzky et al. (1998) | Q | |
| | $1.2\times10^{-3}$ | | Nirmalakhandan and Speece (1988) | Q | |





Table A6.1: Chlorocarbons (C, H, Cl) (. . . continued)

| Substance Formula (Trivial Name) [CAS Registry Number] InChIKey | $H_s^{cp}$ (at $T^\ominus$) $\left[\dfrac{\mathrm{mol}}{\mathrm{m}^3\,\mathrm{Pa}}\right]$ | $\dfrac{\mathrm{d}\ln H_s^{cp}}{\mathrm{d}(1/T)}$ [K] | Reference | Type | Note |
|---|---|---|---|---|---|
| | $3.5\times10^{-3}$ | | Duchowicz et al. (2020) | ? | 185, 21 |
| | $8.5\times10^{-4}$ | | MacBean (2012a) | ? | |
| | $3.5\times10^{-3}$ | | Mackay et al. (2006b) | ? | |
| | | 4000 | Kühne et al. (2005) | ? | |
| | $3.5\times10^{-3}$ | | Yaws (1999) | ? | 21 |
| | $2.2\times10^{-3}$ | | Abraham and Weathersby (1994) | ? | 21 |
| | $3.5\times10^{-3}$ | | Mackay et al. (1993) | ? | |
| | $3.7\times10^{-3}$ | | Yaws and Yang (1992) | ? | 21 |
| | $3.4\times10^{-3}$ | | Abraham et al. (1990) | ? | |
| | $4.8\times10^{-3}$ | | Mackay and Yeun (1983) | ? | |
| | $5.9\times10^{-3}$ | | Chiou et al. (1980) | ? | 79 |
| 1,2-dichloropropane-d6 $C_3D_6Cl_2$ [93952-08-0] KNKRKFALVUDBJE-LIDOUZCJSA-N | $3.6\times10^{-3}$ | 4600 | Hiatt (2013) | M | |
| 1,3-dichloropropane $C_3H_6Cl_2$ [142-28-9] YHRUOJUYPBUZOS-UHFFFAOYSA-N | $1.0\times10^{-2}$ | 3900 | Burkholder et al. (2019) | L | |
| | $1.0\times10^{-2}$ | 3900 | Burkholder et al. (2015) | L | |
| | $1.2\times10^{-2}$ | 3100 | Brockbank (2013) | L | 1 |
| | $1.3\times10^{-2}$ | 5300 | Hiatt (2013) | M | |
| | $1.1\times10^{-2}$ | 5000 | Kondoh and Nakajima (1997) | M | |
| | $8.5\times10^{-3}$ | | Albanese et al. (1987) | M | |
| | $1.0\times10^{-2}$ | 3900 | Leighton and Calo (1981) | M | |
| | $9.9\times10^{-3}$ | | Hine and Mookerjee (1975) | V | |
| | $1.6\times10^{-2}$ | | Keshavarz et al. (2022) | Q | |
| | $3.6\times10^{-3}$ | | Duchowicz et al. (2020) | Q | 299 |
| | $1.8\times10^{-2}$ | | Hilal et al. (2008) | Q | |
| | $5.8\times10^{-3}$ | | Modarresi et al. (2007) | Q | 67 |
| | | 3700 | Kühne et al. (2005) | Q | |
| | $5.6\times10^{-3}$ | | Yaffe et al. (2003) | Q | 248, 249 |
| | $3.0\times10^{-3}$ | | Katritzky et al. (1998) | Q | |
| | $1.4\times10^{-3}$ | | Nirmalakhandan and Speece (1988) | Q | |
| | $1.0\times10^{-2}$ | | Duchowicz et al. (2020) | ? | 185, 21 |
| | | 3900 | Kühne et al. (2005) | ? | |
| | $1.0\times10^{-2}$ | | Yaws (1999) | ? | 21 |
| | $9.9\times10^{-3}$ | | Yaws and Yang (1992) | ? | 21 |
| | $9.9\times10^{-3}$ | | Abraham et al. (1990) | ? | |
| 2,2-dichloropropane $C_3H_6Cl_2$ [594-20-7] ZEOVXNVKXIPWMS-UHFFFAOYSA-N | $4.4\times10^{-4}$ | 7400 | Hiatt (2013) | M | |
| | $8.1\times10^{-4}$ | 3900 | Bakierowska and Trzeszczyński (2003) | M | |
| | $7.1\times10^{-4}$ | 630 | Kondoh and Nakajima (1997) | M | |
| | | 3700 | Kühne et al. (2005) | Q | |
| | | 3900 | Kühne et al. (2005) | ? | |





Table A6.1: Chlorocarbons (C, H, Cl) (...continued)

| Substance<br>Formula<br>(Trivial Name)<br>[CAS Registry Number]<br>InChIKey | $H_s^{cp}$<br>(at $T^\ominus$)<br>$\left[\dfrac{\mathrm{mol}}{\mathrm{m^3\,Pa}}\right]$ | $\dfrac{\mathrm{d}\ln H_s^{cp}}{\mathrm{d}(1/T)}$<br><br>[K] | Reference | Type | Note |
|---|---|---|---|---|---|
| 1,1,1-trichloropropane | $3.1\times10^{-3}$ | | Duchowicz et al. (2020) | V | 186 |
| $C_3H_5Cl_3$ | $3.8\times10^{-3}$ | | Yaws et al. (2005) | X | 446 |
| [7789-89-1] | $1.2\times10^{-3}$ | | Duchowicz et al. (2020) | Q | |
| AVGQTJUPLKNPQP-UHFFFAOYSA-N | $1.1\times10^{-3}$ | | Hilal et al. (2008) | Q | |
| | $3.1\times10^{-3}$ | | Yaffe et al. (2003) | Q | 248, 249 |
| | $6.5\times10^{-4}$ | | Katritzky et al. (1998) | Q | |
| 1,1,2-trichloropropane | $3.1\times10^{-2}$ | | Duchowicz et al. (2020) | V | 186 |
| $C_3H_5Cl_3$ | $1.4\times10^{-2}$ | | Yaws et al. (2005) | X | 446 |
| [598-77-6] | $4.0\times10^{-3}$ | | Duchowicz et al. (2020) | Q | |
| GRSQYISVQKPZCW-UHFFFAOYSA-N | $7.9\times10^{-3}$ | | Hilal et al. (2008) | Q | |
| | $3.1\times10^{-2}$ | | Yaffe et al. (2003) | Q | 248, 249 |
| | $2.4\times10^{-3}$ | | Katritzky et al. (1998) | Q | |
| 1,2,3-trichloropropane | $3.6\times10^{-2}$ | 3700 | Burkholder et al. (2019) | L | |
| $C_3H_5Cl_3$ | $3.6\times10^{-2}$ | 3700 | Burkholder et al. (2015) | L | |
| [96-18-4] | $3.3\times10^{-2}$ | 1900 | Brockbank (2013) | L | 1 |
| CFXQEHVMCRXUSD-UHFFFAOYSA-N | $3.6\times10^{-2}$ | 3700 | Staudinger and Roberts (2001) | L | |
| | $3.4\times10^{-2}$ | 3700 | Staudinger and Roberts (1996) | L | |
| | $4.2\times10^{-2}$ | 7200 | Hiatt (2013) | M | |
| | $2.8\times10^{-2}$ | 5300 | Kondoh and Nakajima (1997) | M | |
| | $4.4\times10^{-2}$ | 4000 | Tancrède and Yanagisawa (1990) | M | |
| | $3.3\times10^{-2}$ | | Albanese et al. (1987) | M | |
| | $2.9\times10^{-2}$ | 3500 | Leighton and Calo (1981) | M | |
| | $2.6\times10^{-2}$ | | Mackay et al. (2006b) | V | |
| | $2.6\times10^{-2}$ | | Mackay et al. (1993) | V | |
| | $3.1\times10^{-2}$ | | Dilling (1977) | V | |
| | $2.6\times10^{-2}$ | | Yaws (2003) | X | 237 |
| | $2.2\times10^{-2}$ | | Yaws et al. (2005) | X | 446 |
| | $1.6\times10^{-2}$ | | Keshavarz et al. (2022) | Q | |
| | $5.3\times10^{-3}$ | | Duchowicz et al. (2020) | Q | |
| | $2.5\times10^{-2}$ | | Raventos-Duran et al. (2010) | Q | 242, 243 |
| | $3.9\times10^{-2}$ | | Raventos-Duran et al. (2010) | Q | 244 |
| | $2.0\times10^{-3}$ | | Raventos-Duran et al. (2010) | Q | 245 |
| | $2.2\times10^{-2}$ | | Gharagheizi et al. (2010) | Q | 246 |
| | $3.9\times10^{-2}$ | | Hilal et al. (2008) | Q | |
| | $1.4\times10^{-2}$ | | Modarresi et al. (2007) | Q | 67 |
| | | 4000 | Kühne et al. (2005) | Q | |
| | $2.9\times10^{-2}$ | | Yaffe et al. (2003) | Q | 248, 249 |
| | $3.9\times10^{-3}$ | | Katritzky et al. (1998) | Q | |
| | $2.9\times10^{-2}$ | | Duchowicz et al. (2020) | ? | 185, 21 |
| | | 4100 | Kühne et al. (2005) | ? | |
| | $2.6\times10^{-2}$ | | Yaws (1999) | ? | 21 |
| | $2.9\times10^{-2}$ | | Yaws and Yang (1992) | ? | 21 |



Table A6.1: Chlorocarbons (C, H, Cl) (. . . continued)

| Substance Formula (Trivial Name) [CAS Registry Number] InChIKey | $H_s^{cp}$ (at $T^{\ominus}$) $\left[\dfrac{\text{mol}}{\text{m}^3\,\text{Pa}}\right]$ | $\dfrac{\text{d}\ln H_s^{cp}}{\text{d}(1/T)}$ [K] | Reference | Type | Note |
|---|---|---|---|---|---|
| 1,1,2,2,3-pentachloropropane | $1.4\times10^{-2}$ | | Zhang et al. (2010) | Q | 287, 288 |
| $C_3H_3Cl_5$ | $7.3\times10^{-2}$ | | Zhang et al. (2010) | Q | 287, 289 |
| [16714-68-4] | $6.2\times10^{-1}$ | | Zhang et al. (2010) | Q | 287, 290 |
| IYFMQUDCYNWFTL-UHFFFAOYSA-N | $8.6\times10^{-3}$ | | Zhang et al. (2010) | Q | 287, 291 |
| 1-chloro-2-methylpropane | $8.3\times10^{-3}$ | | Mackay and Shiu (1981) | L | |
| $C_4H_9Cl$ | $5.0\times10^{-4}$ | | Duchowicz et al. (2020) | V | 186 |
| [513-36-0] | $5.0\times10^{-4}$ | | Duchowicz et al. (2020) | Q | |
| QTBFPMKWQKYFLR-UHFFFAOYSA-N | $3.9\times10^{-4}$ | | Raventos-Duran et al. (2010) | Q | 242, 243 |
| | $6.2\times10^{-4}$ | | Raventos-Duran et al. (2010) | Q | 244 |
| | $4.9\times10^{-4}$ | | Raventos-Duran et al. (2010) | Q | 245 |
| | $7.3\times10^{-4}$ | | Hilal et al. (2008) | Q | |
| | $8.5\times10^{-4}$ | | Modarresi et al. (2007) | Q | 67 |
| | $8.6\times10^{-3}$ | | Yaffe et al. (2003) | Q | 248, 249 |
| | $6.3\times10^{-4}$ | | Yaws and Yang (1992) | ? | 21, 12 |
| 2-chloro-2-methylpropane | $1.6\times10^{-4}$ | 5600 | Clarke et al. (1962) | M | |
| $C_4H_9Cl$ | $7.7\times10^{-4}$ | | Duchowicz et al. (2020) | V | 186 |
| (*tert*-butyl chloride) | $1.2\times10^{-3}$ | | Yaws (2003) | X | 237, 80 |
| [507-20-0] | $2.2\times10^{-4}$ | | Duchowicz et al. (2020) | Q | |
| NBRKLOOSMBRFMH-UHFFFAOYSA-N | $1.0\times10^{-3}$ | | Gharagheizi et al. (2012) | Q | |
| | $3.9\times10^{-4}$ | | Raventos-Duran et al. (2010) | Q | 271, 243 |
| | $2.5\times10^{-4}$ | | Raventos-Duran et al. (2010) | Q | 244 |
| | $4.9\times10^{-4}$ | | Raventos-Duran et al. (2010) | Q | 245 |
| | $1.2\times10^{-3}$ | | Gharagheizi et al. (2010) | Q | 246 |
| | $2.2\times10^{-4}$ | | Hilal et al. (2008) | Q | |
| | $6.5\times10^{-4}$ | | Modarresi et al. (2007) | Q | 67 |
| | $6.9\times10^{-5}$ | | Yaffe et al. (2003) | Q | 248, 249 |
| | $3.1\times10^{-4}$ | | Nirmalakhandan et al. (1997) | Q | |
| | $7.7\times10^{-4}$ | | Yaws (1999) | ? | 21, 80 |
| | $6.4\times10^{-5}$ | | Abraham et al. (1990) | ? | |
| 1-chlorobutane | $5.6\times10^{-4}$ | 4500 | Brockbank (2013) | L | 1 |
| $C_4H_9Cl$ | $6.7\times10^{-4}$ | | Dohnal and Hovorka (1999) | M | 12 |
| (butyl chloride) | $5.3\times10^{-4}$ | | Li et al. (1993) | M | |
| [109-69-3] | $5.9\times10^{-4}$ | 3500 | Leighton and Calo (1981) | M | |
| VFWCMGCRMGJXDK-UHFFFAOYSA-N | $3.3\times10^{-4}$ | | Sato and Nakajima (1979b) | M | 14 |
| | $4.8\times10^{-4}$ | | Mackay et al. (2006b) | V | |
| | $4.8\times10^{-4}$ | | Mackay et al. (1993) | V | |
| | $5.3\times10^{-4}$ | | Abraham (1984) | V | |
| | $5.1\times10^{-4}$ | | Hine and Mookerjee (1975) | V | |
| | $5.9\times10^{-4}$ | | Yaws (2003) | X | 237 |
| | $4.4\times10^{-4}$ | | Keshavarz et al. (2022) | Q | |
| | $1.3\times10^{-3}$ | | Duchowicz et al. (2020) | Q | |
| | $3.1\times10^{-4}$ | | Gharagheizi et al. (2012) | Q | |
| | $3.9\times10^{-4}$ | | Raventos-Duran et al. (2010) | Q | 242, 243 |
| | $6.2\times10^{-4}$ | | Raventos-Duran et al. (2010) | Q | 244 |
| | $4.9\times10^{-4}$ | | Raventos-Duran et al. (2010) | Q | 245 |
| | $4.5\times10^{-4}$ | | Gharagheizi et al. (2010) | Q | 246 |





Table A6.1: Chlorocarbons (C, H, Cl) (... continued)

| Substance Formula (Trivial Name) [CAS Registry Number] InChIKey | $H_s^{cp}$ (at $T^\ominus$) $\left[\dfrac{\text{mol}}{\text{m}^3\,\text{Pa}}\right]$ | $\dfrac{\text{d}\ln H_s^{cp}}{\text{d}(1/T)}$ [K] | Reference | Type | Note |
|---|---|---|---|---|---|
| | $9.0\times10^{-4}$ | | Hilal et al. (2008) | Q | |
| | $8.7\times10^{-4}$ | | Modarresi et al. (2007) | Q | 67 |
| | | 3700 | Kühne et al. (2005) | Q | |
| | $6.1\times10^{-4}$ | | Yaffe et al. (2003) | Q | 248, 249 |
| | $1.9\times10^{-4}$ | | Yao et al. (2002) | Q | 229 |
| | $5.4\times10^{-4}$ | | English and Carroll (2001) | Q | 230, 231 |
| | $1.4\times10^{-3}$ | | Katritzky et al. (1998) | Q | |
| | $1.3\times10^{-3}$ | | Russell et al. (1992) | Q | 279 |
| | $3.6\times10^{-4}$ | | Suzuki et al. (1992) | Q | 232 |
| | $5.0\times10^{-4}$ | | Nirmalakhandan and Speece (1988) | Q | |
| | $5.9\times10^{-4}$ | | Duchowicz et al. (2020) | ? | 185, 21 |
| | $6.5\times10^{-4}$ | | Mackay et al. (2006b) | ? | |
| | | 3700 | Kühne et al. (2005) | ? | |
| | $5.9\times10^{-4}$ | | Yaws (1999) | ? | 21 |
| | $3.4\times10^{-4}$ | | Abraham and Weathersby (1994) | ? | 21 |
| | $6.5\times10^{-4}$ | | Mackay et al. (1993) | ? | |
| | $5.6\times10^{-4}$ | | Hoff et al. (1993) | ? | 21 |
| | $5.8\times10^{-4}$ | | Yaws and Yang (1992) | ? | 21 |
| | $5.3\times10^{-4}$ | | Abraham et al. (1990) | ? | |
| 2-chlorobutane C$_4$H$_9$Cl (*sec*-butyl chloride) [78-86-4] BSPCSKHALVHRSR-UHFFFAOYSA-N | $4.1\times10^{-4}$ | 4500 | Brockbank (2013) | L | 1 |
| | $4.1\times10^{-4}$ | 4500 | Leighton and Calo (1981) | M | |
| | $5.3\times10^{-4}$ | | Mackay et al. (2006b) | V | |
| | $5.3\times10^{-4}$ | | Mackay et al. (1993) | V | |
| | $5.1\times10^{-4}$ | | Yaws (2003) | X | 237 |
| | $4.4\times10^{-4}$ | | Keshavarz et al. (2022) | Q | |
| | $5.9\times10^{-4}$ | | Duchowicz et al. (2020) | Q | 299 |
| | $5.3\times10^{-4}$ | | Gharagheizi et al. (2012) | Q | |
| | $3.9\times10^{-4}$ | | Raventos-Duran et al. (2010) | Q | 242, 243 |
| | $4.9\times10^{-4}$ | | Raventos-Duran et al. (2010) | Q | 244 |
| | $4.9\times10^{-4}$ | | Raventos-Duran et al. (2010) | Q | 245 |
| | $3.7\times10^{-4}$ | | Gharagheizi et al. (2010) | Q | 246 |
| | $6.2\times10^{-4}$ | | Hilal et al. (2008) | Q | |
| | $8.2\times10^{-4}$ | | Modarresi et al. (2007) | Q | 67 |
| | | 3700 | Kühne et al. (2005) | Q | |
| | $4.2\times10^{-4}$ | | Yaffe et al. (2003) | Q | 248, 249 |
| | $1.3\times10^{-4}$ | | Yao et al. (2002) | Q | 229 |
| | $4.7\times10^{-4}$ | | English and Carroll (2001) | Q | 230, 274 |
| | $1.9\times10^{-3}$ | | Katritzky et al. (1998) | Q | |
| | $4.2\times10^{-4}$ | | Nirmalakhandan et al. (1997) | Q | |
| | $4.1\times10^{-4}$ | | Duchowicz et al. (2020) | ? | 185, 21 |
| | $4.4\times10^{-4}$ | | Mackay et al. (2006b) | ? | |
| | | 4500 | Kühne et al. (2005) | ? | |
| | $5.2\times10^{-4}$ | | Yaws (1999) | ? | 21 |
| | $4.4\times10^{-4}$ | | Mackay et al. (1993) | ? | |
| | $5.3\times10^{-4}$ | | Yaws and Yang (1992) | ? | 21 |
| | $4.0\times10^{-4}$ | | Abraham et al. (1990) | ? | |



Table A6.1: Chlorocarbons (C, H, Cl) (. . . continued)

| Substance Formula (Trivial Name) [CAS Registry Number] InChIKey | $H_s^{cp}$ (at $T^\ominus$) $\left[\dfrac{\text{mol}}{\text{m}^3\,\text{Pa}}\right]$ | $\dfrac{\text{d}\ln H_s^{cp}}{\text{d}(1/T)}$ [K] | Reference | Type | Note |
|---|---|---|---|---|---|
| 1,1-dichlorobutane C$_4$H$_8$Cl$_2$ [541-33-3] SEQRDAAUNCRFIT-UHFFFAOYSA-N | $1.3\times10^{-3}$ | | Duchowicz et al. (2020) | V | 186 |
| | $1.3\times10^{-3}$ | | Hine and Mookerjee (1975) | V | |
| | $2.7\times10^{-3}$ | | Duchowicz et al. (2020) | Q | |
| | $2.5\times10^{-3}$ | | Hilal et al. (2008) | Q | |
| | $9.2\times10^{-4}$ | | Modarresi et al. (2007) | Q | 67 |
| | $1.4\times10^{-3}$ | | Yaffe et al. (2003) | Q | 248, 249 |
| | $1.6\times10^{-3}$ | | English and Carroll (2001) | Q | 230, 260 |
| | $1.3\times10^{-3}$ | | Katritzky et al. (1998) | Q | |
| | $9.2\times10^{-4}$ | | Nirmalakhandan and Speece (1988) | Q | |
| 1,4-dichlorobutane C$_4$H$_8$Cl$_2$ [110-56-5] KJDRSWPQXHESDQ-UHFFFAOYSA-N | $2.0\times10^{-2}$ | 3600 | Brockbank (2013) | L | 1 |
| | $1.2\times10^{-2}$ | | Albanese et al. (1987) | M | |
| | $2.0\times10^{-2}$ | 3100 | Leighton and Calo (1981) | M | |
| | $2.2\times10^{-2}$ | | Keshavarz et al. (2022) | Q | |
| | $3.6\times10^{-3}$ | | Duchowicz et al. (2020) | Q | 299 |
| | $2.6\times10^{-2}$ | | Hilal et al. (2008) | Q | |
| | $4.5\times10^{-3}$ | | Modarresi et al. (2007) | Q | 67 |
| | | 4000 | Kühne et al. (2005) | Q | |
| | $2.0\times10^{-2}$ | | Yaffe et al. (2003) | Q | 248, 249 |
| | $1.9\times10^{-3}$ | | English and Carroll (2001) | Q | 230, 231 |
| | $3.9\times10^{-3}$ | | Katritzky et al. (1998) | Q | |
| | $1.1\times10^{-3}$ | | Nirmalakhandan et al. (1997) | Q | |
| | $2.0\times10^{-2}$ | | Duchowicz et al. (2020) | ? | 185, 21 |
| | | 3700 | Kühne et al. (2005) | ? | |
| | $2.0\times10^{-2}$ | | Abraham et al. (1990) | ? | |
| 2,3-dichlorobutane C$_4$H$_8$Cl$_2$ [7581-97-7] RMISVOPUIFJTEO-UHFFFAOYSA-N | $1.4\times10^{-3}$ | | Duchowicz et al. (2020) | V | 186 |
| | $2.5\times10^{-3}$ | | Yaws et al. (2005) | X | 446 |
| | $8.4\times10^{-4}$ | | Duchowicz et al. (2020) | Q | |
| | $2.8\times10^{-3}$ | | Hilal et al. (2008) | Q | |
| | $2.5\times10^{-3}$ | | Modarresi et al. (2007) | Q | 67 |
| | $1.4\times10^{-3}$ | | Yaffe et al. (2003) | Q | 248, 249 |
| | $3.1\times10^{-3}$ | | Katritzky et al. (1998) | Q | |
| 1-chloropentane C$_5$H$_{11}$Cl [543-59-9] SQCZQTSHSZLZIQ-UHFFFAOYSA-N | $4.1\times10^{-4}$ | 4000 | Brockbank (2013) | L | 1 |
| | $4.2\times10^{-4}$ | | Li et al. (1993) | M | |
| | $4.1\times10^{-4}$ | 4700 | Leighton and Calo (1981) | M | |
| | $2.7\times10^{-4}$ | | Sato and Nakajima (1979b) | M | 14 |
| | $4.5\times10^{-4}$ | | Mackay et al. (2006b) | V | |
| | $4.5\times10^{-4}$ | | Mackay et al. (1993) | V | |
| | $4.5\times10^{-4}$ | | Abraham (1984) | V | |
| | $4.3\times10^{-4}$ | | Amoore and Buttery (1978) | V | |
| | $4.5\times10^{-4}$ | | Hine and Mookerjee (1975) | V | |
| | $1.9\times10^{-4}$ | | Yaws (2003) | X | 237 |
| | $5.9\times10^{-4}$ | | Keshavarz et al. (2022) | Q | |
| | $1.3\times10^{-3}$ | | Duchowicz et al. (2020) | Q | 299 |
| | $2.5\times10^{-4}$ | | Gharagheizi et al. (2012) | Q | |
| | $2.9\times10^{-4}$ | | Gharagheizi et al. (2010) | Q | 246 |
| | $7.3\times10^{-4}$ | | Hilal et al. (2008) | Q | |





Table A6.1: Chlorocarbons (C, H, Cl) (...continued)

| Substance<br>Formula<br>(Trivial Name)<br>[CAS Registry Number]<br>InChIKey | $H_s^{cp}$<br>(at $T^{\ominus}$)<br>$\left[\dfrac{\mathrm{mol}}{\mathrm{m^3\,Pa}}\right]$ | $\dfrac{\mathrm{d}\ln H_s^{cp}}{\mathrm{d}(1/T)}$<br><br>[K] | Reference | Type | Note |
|---|---|---|---|---|---|
| | $7.1\times10^{-4}$ | | Modarresi et al. (2007) | Q | 67 |
| | | 4000 | Kühne et al. (2005) | Q | |
| | $4.2\times10^{-4}$ | | Yaffe et al. (2003) | Q | 248, 249 |
| | $4.1\times10^{-4}$ | | English and Carroll (2001) | Q | 230, 231 |
| | $1.6\times10^{-3}$ | | Katritzky et al. (1998) | Q | |
| | $2.8\times10^{-4}$ | | Suzuki et al. (1992) | Q | 232 |
| | $3.9\times10^{-4}$ | | Nirmalakhandan and Speece (1988) | Q | |
| | $4.1\times10^{-4}$ | | Duchowicz et al. (2020) | ? | 185, 21 |
| | $4.2\times10^{-4}$ | | Mackay et al. (2006b) | ? | |
| | | 4400 | Kühne et al. (2005) | ? | |
| | $1.9\times10^{-4}$ | | Yaws (1999) | ? | 21 |
| | $2.8\times10^{-4}$ | | Abraham and Weathersby (1994) | ? | 21 |
| | $4.2\times10^{-4}$ | | Mackay et al. (1993) | ? | |
| | $2.0\times10^{-4}$ | | Yaws and Yang (1992) | ? | 21 |
| | $4.5\times10^{-4}$ | | Abraham et al. (1990) | ? | |
| 2-chloropentane<br>$C_5H_{11}Cl$<br>[625-29-6]<br>NFRKUDYZEVQXTE-UHFFFAOYSA-N | $6.7\times10^{-4}$ | | Duchowicz et al. (2020) | V | 186 |
| | $3.6\times10^{-4}$ | | Hine and Mookerjee (1975) | V | |
| | $5.8\times10^{-4}$ | | Duchowicz et al. (2020) | Q | |
| | $4.8\times10^{-4}$ | | Hilal et al. (2008) | Q | |
| | $6.0\times10^{-4}$ | | Modarresi et al. (2007) | Q | 67 |
| | $6.7\times10^{-4}$ | | Yaffe et al. (2003) | Q | 248, 249 |
| | $3.6\times10^{-4}$ | | English and Carroll (2001) | Q | 230, 274 |
| | $2.2\times10^{-3}$ | | Katritzky et al. (1998) | Q | |
| | $2.5\times10^{-4}$ | | Suzuki et al. (1992) | Q | 232 |
| | $3.3\times10^{-4}$ | | Nirmalakhandan and Speece (1988) | Q | |
| | $3.6\times10^{-4}$ | | Abraham et al. (1990) | ? | |
| 3-chloropentane<br>$C_5H_{11}Cl$<br>[616-20-6]<br>CXQSCYIVCSCSES-UHFFFAOYSA-N | $3.8\times10^{-4}$ | | Duchowicz et al. (2020) | V | 186 |
| | $3.8\times10^{-4}$ | | Meylan and Howard (1991) | V | |
| | $3.8\times10^{-4}$ | | Hine and Mookerjee (1975) | V | |
| | $5.8\times10^{-4}$ | | Duchowicz et al. (2020) | Q | |
| | $4.7\times10^{-4}$ | | Hilal et al. (2008) | Q | |
| | $5.9\times10^{-4}$ | | Modarresi et al. (2007) | Q | 67 |
| | $4.5\times10^{-4}$ | | English and Carroll (2001) | Q | 230, 260 |
| | $2.5\times10^{-4}$ | | Suzuki et al. (1992) | Q | 232 |
| | $3.9\times10^{-4}$ | | Meylan and Howard (1991) | Q | |
| | $3.4\times10^{-4}$ | | Nirmalakhandan and Speece (1988) | Q | |
| | $3.8\times10^{-4}$ | | Abraham et al. (1990) | ? | |
| 1,2-dichloropentane<br>$C_5H_{10}Cl_2$<br>[1674-33-5]<br>PPLBPDUKNRCHGG-UHFFFAOYSA-N | $4.8\times10^{-3}$ | | Yaws et al. (2005) | X | 446 |
| | $3.1\times10^{-3}$ | | Hilal et al. (2008) | Q | |



Table A6.1: Chlorocarbons (C, H, Cl) (. . . continued)

| Substance Formula (Trivial Name) [CAS Registry Number] InChIKey | $H_s^{cp}$ (at $T^\ominus$) $\left[\dfrac{\text{mol}}{\text{m}^3\,\text{Pa}}\right]$ | $\dfrac{\text{d}\ln H_s^{cp}}{\text{d}(1/T)}$ [K] | Reference | Type | Note |
|---|---|---|---|---|---|
| 1,5-dichloropentane $C_5H_{10}Cl_2$ [628-76-2] LBKDGROORAKTLC-UHFFFAOYSA-N | $1.0\times10^{-2}$ $1.8\times10^{-2}$ $2.9\times10^{-2}$ $3.6\times10^{-3}$ $2.0\times10^{-2}$ $3.8\times10^{-3}$  $1.8\times10^{-2}$ $4.2\times10^{-3}$ $1.8\times10^{-2}$ | 4000 1600    4400   4100 | Brockbank (2013) Leighton and Calo (1981) Keshavarz et al. (2022) Duchowicz et al. (2020) Hilal et al. (2008) Modarresi et al. (2007) Kühne et al. (2005) Yaffe et al. (2003) Katritzky et al. (1998) Duchowicz et al. (2020) Kühne et al. (2005) | L M Q Q Q Q Q Q Q ? ? | 1   184  67  248, 249  185, 21 |
| 2,3-dichloropentane $C_5H_{10}Cl_2$ [600-11-3] HVFJQRZGBBKTPL-UHFFFAOYSA-N | $2.9\times10^{-3}$ $2.8\times10^{-3}$ | | Yaws et al. (2005) Hilal et al. (2008) | X Q | 446 |
| 2-chloro-2-methylbutane $C_5H_{11}Cl$ [594-36-5] CRNIHJHMEQZAAS-UHFFFAOYSA-N | $3.1\times10^{-3}$ $3.0\times10^{-3}$ | | Yaws (1999) Yaws and Yang (1992) | ? ? | 21 21 |
| 1-chlorohexane $C_6H_{13}Cl$ [544-10-5] MLRVZFYXUZQSRU-UHFFFAOYSA-N | $3.1\times10^{-4}$ $4.1\times10^{-4}$ $8.0\times10^{-4}$ $1.3\times10^{-3}$ $6.1\times10^{-4}$ $6.1\times10^{-4}$  $3.5\times10^{-4}$ $3.1\times10^{-4}$ $1.7\times10^{-3}$ $3.1\times10^{-4}$ $4.1\times10^{-4}$  $4.0\times10^{-4}$ | 4500    4300    4400 | Li et al. (1993) Leighton and Calo (1981) Keshavarz et al. (2022) Duchowicz et al. (2020) Hilal et al. (2008) Modarresi et al. (2007) Kühne et al. (2005) Yaffe et al. (2003) English and Carroll (2001) Katritzky et al. (1998) Nirmalakhandan et al. (1997) Duchowicz et al. (2020) Kühne et al. (2005) Abraham et al. (1990) | M M Q Q Q Q Q Q Q Q Q ? ? ? | 299  67  248, 249 230, 231  185, 21 |
| 2-chlorohexane $C_6H_{13}Cl$ [638-28-8] GLCIPJOIEVLTPR-UHFFFAOYSA-N | $5.0\times10^{-4}$ $4.2\times10^{-4}$ | | Yaws et al. (2005) Hilal et al. (2008) | X Q | 446 |
| 3-chlorohexane $C_6H_{13}Cl$ [2346-81-8] BXSMMAVTEURRGG-UHFFFAOYSA-N | $5.0\times10^{-4}$ $5.0\times10^{-4}$ | | Yaws et al. (2005) Hilal et al. (2008) | X Q | 446 |



Table A6.1: Chlorocarbons (C, H, Cl) (...continued)

| Substance Formula (Trivial Name) [CAS Registry Number] InChIKey | $H_s^{cp}$ (at $T^{\ominus}$) $\left[\dfrac{\mathrm{mol}}{\mathrm{m}^3\,\mathrm{Pa}}\right]$ | $\dfrac{\mathrm{d}\ln H_s^{cp}}{\mathrm{d}(1/T)}$ [K] | Reference | Type | Note |
|---|---|---|---|---|---|
| 1-chloroheptane $C_7H_{15}Cl$ [629-06-1] DZMDPHNGKBEVRE-UHFFFAOYSA-N | $2.5\times10^{-4}$ $5.1\times10^{-4}$ $5.1\times10^{-4}$ $2.3\times10^{-4}$ $2.4\times10^{-4}$ $2.5\times10^{-4}$ | | Abraham (1984) Hilal et al. (2008) Modarresi et al. (2007) English and Carroll (2001) Nirmalakhandan et al. (1997) Abraham et al. (1990) | V Q Q Q Q ? | 67 230, 231 |
| 2-chloroheptane $C_7H_{15}Cl$ [1001-89-4] PTSLUOSUHFGQHV-UHFFFAOYSA-N | $3.9\times10^{-4}$ $3.4\times10^{-4}$ | | Yaws et al. (2005) Hilal et al. (2008) | X Q | 446 |
| 3-chloroheptane $C_7H_{15}Cl$ [999-52-0] DMKNOEJJJSHSML-UHFFFAOYSA-N | $3.6\times10^{-4}$ $3.4\times10^{-4}$ | | Yaws et al. (2005) Hilal et al. (2008) | X Q | 446 |
| 4-chloroheptane $C_7H_{15}Cl$ [998-95-8] MGSGWAXIEMEWCQ-UHFFFAOYSA-N | $3.6\times10^{-4}$ $3.5\times10^{-4}$ | | Yaws et al. (2005) Hilal et al. (2008) | X Q | 446 |
| 1-chlorooctane $C_8H_{17}Cl$ [111-85-3] CNDHHGUSRIZDSL-UHFFFAOYSA-N | $2.6\times10^{-4}$ $1.9\times10^{-4}$ $2.6\times10^{-4}$ $1.3\times10^{-3}$ $1.6\times10^{-4}$ $4.2\times10^{-4}$ $2.6\times10^{-4}$ $1.8\times10^{-3}$ | 6100 | Duchowicz et al. (2020) Sarraute et al. (2004) Yaws et al. (2005) Duchowicz et al. (2020) HSDB (2015) Hilal et al. (2008) Yaffe et al. (2003) Katritzky et al. (1998) | V V X Q Q Q Q Q | 186 446 99 248, 249 |
| 2-chlorooctane $C_8H_{17}Cl$ [628-61-5] HKDCIIMOALDWHF-UHFFFAOYSA-N | $2.7\times10^{-4}$ $3.1\times10^{-4}$ | | Yaws et al. (2005) Hilal et al. (2008) | X Q | 446 |
| 3-(chloromethyl)-heptane $C_8H_{17}Cl$ [123-04-6] WLVCBAMXYMWGLJ-UHFFFAOYSA-N | $4.5\times10^{-4}$ $4.9\times10^{-4}$ | | Hilal et al. (2008) Modarresi et al. (2007) | Q Q | 67 |
| 1,8-dichlorooctane $C_8H_{16}Cl_2$ [2162-99-4] WXYMNDFVLNUAIA-UHFFFAOYSA-N | $7.1\times10^{-3}$ | 7100 | Sarraute et al. (2006) | M | 678 |
| 1-chlorononane $C_9H_{19}Cl$ [2473-01-0] RKAMCQVGHFRILV-UHFFFAOYSA-N | $1.6\times10^{-4}$ $3.5\times10^{-4}$ | | Yaws et al. (2005) Hilal et al. (2008) | X Q | 446 |



Table A6.1: Chlorocarbons (C, H, Cl) (... continued)

| Substance Formula (Trivial Name) [CAS Registry Number] InChIKey | $H_s^{cp}$ (at $T^{\ominus}$) $\left[\dfrac{\text{mol}}{\text{m}^3\,\text{Pa}}\right]$ | $\dfrac{\mathrm{d}\ln H_s^{cp}}{\mathrm{d}(1/T)}$ [K] | Reference | Type | Note |
|---|---|---|---|---|---|
| 2-chlorononane C$_9$H$_{19}$Cl [2216-36-6] DTWJISBCMBWFNY-UHFFFAOYSA-N | $2.7\times10^{-4}$ $3.0\times10^{-4}$ | | Yaws et al. (2005) Hilal et al. (2008) | X Q | 446 |
| 5-chlorononane C$_9$H$_{19}$Cl [28123-70-8] GHLDSOWZIOPMTC-UHFFFAOYSA-N | $2.2\times10^{-4}$ $2.6\times10^{-4}$ | | Yaws et al. (2005) Hilal et al. (2008) | X Q | 446 |
| 1-chlorodecane C$_{10}$H$_{21}$Cl [1002-69-3] ZTEHOZMYMCEYRM-UHFFFAOYSA-N | $1.6\times10^{-4}$ $2.5\times10^{-4}$ | | Yaws et al. (2005) Hilal et al. (2008) | X Q | 446 |
| 1,10-dichlorodecane C$_{10}$H$_{20}$Cl$_2$ [2162-98-3] RBBNTRDPSVZESY-UHFFFAOYSA-N | $2.0\times10^{-3}$ $5.3\times10^{-3}$ | | Drouillard et al. (1998) Hilal et al. (2008) | V Q | |
| 1,2,9,10-tetrachlorodecane C$_{10}$H$_{18}$Cl$_4$ [205646-11-3] VXBHNYIEBLRXAW-UHFFFAOYSA-N | $5.6\times10^{-2}$ $6.2\times10^{-2}$ $3.9\times10^{-2}$ $6.2\times10^{-4}$ $1.4\times10^{-2}$ | | Drouillard et al. (1998) Raventos-Duran et al. (2010) Raventos-Duran et al. (2010) Raventos-Duran et al. (2010) Hilal et al. (2008) | M Q Q Q Q | 242, 243 244 245 |
| pentachlorodecane isomers C$_{10}$H$_{17}$Cl$_5$ [175801-37-3] AMKBEJYNNQNKGD-UHFFFAOYSA-N | $2.0\times10^{-1}$ $3.8\times10^{-1}$ | | Drouillard et al. (1998) Drouillard et al. (1998) | M M | |
| 1-chloroundecane C$_{11}$H$_{23}$Cl [2473-03-2] ZHKKNUKCXPWZOP-UHFFFAOYSA-N | $1.7\times10^{-4}$ $2.3\times10^{-4}$ | | Yaws et al. (2005) Hilal et al. (2008) | X Q | 446 |
| 1,2,10,11-tetrachloroundecane C$_{11}$H$_{20}$Cl$_4$ [210049-49-3] VVAAXDBMZVCVPA-UHFFFAOYSA-N | $1.6\times10^{-1}$ $4.9\times10^{-2}$ $3.1\times10^{-2}$ $4.9\times10^{-4}$ $1.1\times10^{-2}$ | | Drouillard et al. (1998) Raventos-Duran et al. (2010) Raventos-Duran et al. (2010) Raventos-Duran et al. (2010) Hilal et al. (2008) | M Q Q Q Q | 242, 243 244 245 |
| pentachloroundecane isomers C$_{11}$H$_{19}$Cl$_5$ [210175-48-7] BBCUCNDIXLWBNQ-UHFFFAOYSA-N | $6.8\times10^{-1}$ $1.5$ | | Drouillard et al. (1998) Drouillard et al. (1998) | M M | |
| 1-chlorododecane C$_{12}$H$_{25}$Cl [112-52-7] YAYNEUUHHLGGAH-UHFFFAOYSA-N | $2.3\times10^{-4}$ $1.9\times10^{-4}$ | | Yaws et al. (2005) Hilal et al. (2008) | X Q | 446 |





Table A6.1: Chlorocarbons (C, H, Cl) (...continued)

| Substance<br>Formula<br>(Trivial Name)<br>[CAS Registry Number]<br>InChIKey | $H_s^{cp}$ (at $T^\ominus$) $\left[\dfrac{\text{mol}}{\text{m}^3\,\text{Pa}}\right]$ | $\dfrac{\text{d}\ln H_s^{cp}}{\text{d}(1/T)}$ [K] | Reference | Type | Note |
|---|---|---|---|---|---|
| 1,12-dichlorododecane<br>$C_{12}H_{24}Cl_2$<br>[3922-28-9]<br>RNXPZVYZVHJVHM-UHFFFAOYSA-N | $1.5\times10^{-3}$<br>$3.1\times10^{-3}$ | | Drouillard et al. (1998)<br>Hilal et al. (2008) | V<br>Q | |
| 1-chlorotridecane<br>$C_{13}H_{27}Cl$<br>[822-13-9]<br>ASZMYJSJEOGSBR-UHFFFAOYSA-N | $2.9\times10^{-4}$<br>$1.4\times10^{-4}$ | | Yaws et al. (2005)<br>Hilal et al. (2008) | X<br>Q | 446 |
| 1-chlorotetradecane<br>$C_{14}H_{29}Cl$<br>[2425-54-9]<br>RNHWYOLIEJIAMV-UHFFFAOYSA-N | $3.9\times10^{-4}$<br>$1.2\times10^{-4}$ | | Yaws et al. (2005)<br>Hilal et al. (2008) | X<br>Q | 446 |
| tetrachlorocyclopentane<br>$C_5H_6Cl_4$<br>[59808-78-5]<br>ZFMWDTNZPKDVBU-UHFFFAOYSA-N | $6.4\times10^{-3}$<br>$4.1\times10^{-1}$<br>$1.5$<br>$2.9\times10^{-2}$ | | Zhang et al. (2010)<br>Zhang et al. (2010)<br>Zhang et al. (2010)<br>Zhang et al. (2010) | Q<br>Q<br>Q<br>Q | 287, 288<br>287, 289<br>287, 290<br>287, 291 |
| 1,1,2,3,3,4-<br>hexachlorocyclopentane<br>$C_5H_4Cl_6$<br>[68258-91-3]<br>RPUFWOAXMFQSDJ-UHFFFAOYSA-N | $5.1\times10^{-2}$<br><br>$1.9\times10^{-1}$<br>$1.6$<br>$2.2\times10^{-2}$ | | Zhang et al. (2010)<br><br>Zhang et al. (2010)<br>Zhang et al. (2010)<br>Zhang et al. (2010) | Q<br><br>Q<br>Q<br>Q | 287, 288<br><br>287, 289<br>287, 290<br>287, 291 |
| 1,1,2,3,3,4,5-<br>heptachlorocyclopentane<br>$C_5H_3Cl_7$<br>[68258-90-2]<br>XCEUTYGYMGYCBG-UHFFFAOYSA-N | $1.5\times10^{-1}$<br><br>$7.9\times10^{-1}$<br>$1.6$<br>$8.6\times10^{-2}$ | | Zhang et al. (2010)<br><br>Zhang et al. (2010)<br>Zhang et al. (2010)<br>Zhang et al. (2010) | Q<br><br>Q<br>Q<br>Q | 287, 288<br><br>287, 289<br>287, 290<br>287, 291 |
| 1,2,3,3,4,5-<br>hexachlorocyclopentene<br>$C_5H_2Cl_6$<br>OSRFTCIVMWPVNP-UHFFFAOYSA-N | $1.4\times10^{-2}$<br><br>$4.4\times10^{-2}$<br>$4.4\times10^{-1}$<br>$6.4\times10^{-2}$ | | Zhang et al. (2010)<br><br>Zhang et al. (2010)<br>Zhang et al. (2010)<br>Zhang et al. (2010) | Q<br><br>Q<br>Q<br>Q | 287, 288<br><br>287, 289<br>287, 290<br>287, 291 |
| heptachlorocyclopentene<br>$C_5HCl_7$<br>[62111-47-1]<br>AJUXFTIOMRYFRL-UHFFFAOYSA-N | $3.9\times10^{-2}$<br>$3.5\times10^{-2}$<br>$8.4\times10^{-2}$<br>$5.4\times10^{-2}$ | | Zhang et al. (2010)<br>Zhang et al. (2010)<br>Zhang et al. (2010)<br>Zhang et al. (2010) | Q<br>Q<br>Q<br>Q | 287, 288<br>287, 289<br>287, 290<br>287, 291 |
| chlorocyclohexane<br><br>$C_6H_{11}Cl$<br>[542-18-7]<br>UNFUYWDGSFDHCW-UHFFFAOYSA-N | $2.8\times10^{-3}$<br><br>$3.2\times10^{-3}$ | 3300<br><br><br>4200<br>3200 | Bakierowska and Trzeszczyński (2003)<br>Hilal et al. (2008)<br>Kühne et al. (2005)<br>Kühne et al. (2005) | M<br><br>Q<br>Q<br>? | |




Table A6.1: Chlorocarbons (C, H, Cl) (... continued)

| Substance Formula (Trivial Name) [CAS Registry Number] InChIKey | $H_s^{cp}$ (at $T^{\ominus}$) $\left[\dfrac{\text{mol}}{\text{m}^3\,\text{Pa}}\right]$ | $\dfrac{\text{d}\ln H_s^{cp}}{\text{d}(1/T)}$ [K] | Reference | Type | Note |
|---|---|---|---|---|---|
| $\alpha$-1,2,3,4,5,6-hexachlorocyclohexane | 1.5 | | Xiao et al. (2004) | L | 366 |
| $C_6H_6Cl_6$ | 1.4 | | Xiao et al. (2004) | L | 367 |
| ($\alpha$-lindane; $\alpha$-HCH) | 3.0 | 5500 | Cetin et al. (2006) | M | |
| [319-84-6] | 1.7 | 7500 | Sahsuvar et al. (2003) | M | |
| JLYXXMFPNIAWKQ-SHFUYGGZSA-N | $8.1\times10^{-1}$ | | Altschuh et al. (1999) | M | |
| | 1.3 | 6500 | Kucklick et al. (1991) | M | |
| | $4.2\times10^{-1}$ | | Atlas et al. (1982) | M | 679 |
| | 1.1 | | Mackay et al. (2006d) | V | |
| | $9.1\times10^{-1}$ | | Ballschmiter and Wittlinger (1991) | V | |
| | 2.3 | | Calamari et al. (1991) | V | 12 |
| | 1.1 | | Suntio et al. (1988) | V | 12 |
| | $5.9\times10^{-3}$ | 3900 | Paasivirta et al. (1999) | T | |
| | 1.8 | | Suntio et al. (1988) | C | 680 |
| | $3.9\times10^{-2}$ | | Zhang et al. (2010) | Q | 287, 288 |
| | 7.7 | | Zhang et al. (2010) | Q | 287, 289 |
| | $4.0\times10^{1}$ | | Zhang et al. (2010) | Q | 287, 290 |
| | $3.8\times10^{-1}$ | | Zhang et al. (2010) | Q | 287, 291 |
| | 1.1 | | Modarresi et al. (2007) | Q | 67 |
| | | 7100 | Kühne et al. (2005) | Q | |
| | | 7100 | Kühne et al. (2005) | ? | |
| $\beta$-1,2,3,4,5,6-hexachlorocyclohexane | $2.7\times10^{1}$ | | Xiao et al. (2004) | L | 366 |
| $C_6H_6Cl_6$ | $2.7\times10^{1}$ | | Xiao et al. (2004) | L | 367 |
| ($\beta$-lindane; $\beta$-HCH) | $2.8\times10^{1}$ | 7800 | Sahsuvar et al. (2003) | M | |
| [319-85-7] | $2.2\times10^{1}$ | | Altschuh et al. (1999) | M | |
| JLYXXMFPNIAWKQ-CDRYSYESSA-N | 8.6 | | Mackay et al. (2006d) | V | |
| | $1.4\times10^{1}$ | | Ballschmiter and Wittlinger (1991) | V | |
| | 8.3 | | Suntio et al. (1988) | V | 12 |
| | $5.6\times10^{1}$ | | Suntio et al. (1988) | C | 681 |
| | $6.7\times10^{-1}$ | | Ryan et al. (1988) | C | |
| | 5.8 | | Keshavarz et al. (2022) | Q | |
| | $3.0\times10^{-2}$ | | Duchowicz et al. (2020) | Q | 184 |
| | 1.1 | | Modarresi et al. (2007) | Q | 67 |
| | | 7100 | Kühne et al. (2005) | Q | |
| | $2.2\times10^{1}$ | | Duchowicz et al. (2020) | ? | 185, 21 |
| | | 7800 | Kühne et al. (2005) | ? | |
| $\gamma$-1,2,3,4,5,6-hexachlorocyclohexane | 3.7 | | Xiao et al. (2004) | L | 366 |
| $C_6H_6Cl_6$ | 3.3 | | Xiao et al. (2004) | L | 367 |
| ($\gamma$-lindane; lindane; $\gamma$-HCH) | 3.1 | | Mackay and Shiu (1981) | L | |
| [58-89-9] | 3.1 | | Chao et al. (2017) | M | |
| JLYXXMFPNIAWKQ-GNIYUCBRSA-N | 3.9 | 3300 | Cetin et al. (2006) | M | |
| | 6.0 | 6200 | Xie et al. (2004) | M | |
| | 4.3 | 7500 | Sahsuvar et al. (2003) | M | |
| | 1.9 | | Altschuh et al. (1999) | M | |





Table A6.1: Chlorocarbons (C, H, Cl) (...continued)

| Substance<br>Formula<br>(Trivial Name)<br>[CAS Registry Number]<br>InChIKey | $H_s^{cp}$<br>(at $T^\ominus$)<br>$\left[\dfrac{\text{mol}}{\text{m}^3\,\text{Pa}}\right]$ | $\dfrac{\text{d}\ln H_s^{cp}}{\text{d}(1/T)}$<br><br>[K] | Reference | Type | Note |
|---|---|---|---|---|---|
| | 2.8 | 5500 | Kucklick et al. (1991) | M | |
| | 4.9 | | Fendinger et al. (1989) | M | 72 |
| | 5.0 | | Fendinger and Glotfelty (1988) | M | 72 |
| | 6.7 | | Mackay et al. (2006d) | V | |
| | 3.3 | | Siebers et al. (1994) | V | |
| | $1.0\times10^1$ | | Ballschmiter and Wittlinger (1991) | V | |
| | 5.9 | | Calamari et al. (1991) | V | 12 |
| | 3.7 | | McLachlan et al. (1990) | V | 373 |
| | 7.7 | | Suntio et al. (1988) | V | 12 |
| | $6.7\times10^{-1}$ | | Caron et al. (1985) | V | |
| | 7.9 | | Burkhard and Guth (1981) | V | |
| | 3.1 | | Chiou et al. (1980) | V | |
| | $2.0\times10^1$ | | Mackay and Leinonen (1975) | V | |
| | $6.2\times10^{-2}$ | 7100 | Paasivirta et al. (1999) | T | |
| | $3.1\times10^1$ | | McCarty (1980) | X | 368 |
| | $2.0\times10^1$ | | Suntio et al. (1988) | C | 12 |
| | 5.0 | | Suntio et al. (1988) | C | 681 |
| | 1.4 | | Suntio et al. (1988) | C | |
| | 5.8 | | Keshavarz et al. (2022) | Q | |
| | $3.0\times10^{-2}$ | | Duchowicz et al. (2020) | Q | |
| | $3.9\times10^{-2}$ | | Zhang et al. (2010) | Q | 287, 288 |
| | 7.7 | | Zhang et al. (2010) | Q | 287, 289 |
| | $4.7\times10^1$ | | Zhang et al. (2010) | Q | 287, 290 |
| | $3.8\times10^{-1}$ | | Zhang et al. (2010) | Q | 287, 291 |
| | 5.3 | | Hilal et al. (2008) | Q | |
| | 1.1 | | Modarresi et al. (2007) | Q | 67 |
| | | 7100 | Kühne et al. (2005) | Q | |
| | 1.9 | | Duchowicz et al. (2020) | ? | 185, 21 |
| | | 6200 | Kühne et al. (2005) | ? | |
| | $2.2\times10^1$ | | Brimblecombe (1986) | ? | 80 |
| $\delta$-1,2,3,4,5,6-<br>hexachlorocyclohexane<br>$C_6H_6Cl_6$<br>($\delta$-lindane; $\delta$-HCH)<br>[319-86-8]<br>JLYXXMFPNIAWKQ-GPIVLXJGSA-N | $2.3\times10^1$ | | Duchowicz et al. (2020) | V | 186 |
| | $2.3\times10^1$ | | HSDB (2015) | V | |
| | $1.4\times10^1$ | | Mackay et al. (2006d) | V | |
| | $1.4\times10^1$ | | Suntio et al. (1988) | V | 12 |
| | $5.6\times10^1$ | | Suntio et al. (1988) | C | 681 |
| | $3.0\times10^{-2}$ | | Duchowicz et al. (2020) | Q | |
| | 1.1 | | Modarresi et al. (2007) | Q | 67 |
| 4,5,6,7,8,8-hexachloro-3a,4,7,7a-<br>tetrahydro-4,7-methano-1H-indene<br>$C_{10}H_6Cl_6$<br>[3734-48-3]<br>XCJXQCUJXDUNDN-UHFFFAOYSA-N | $2.0\times10^{-2}$ | | HSDB (2015) | Q | 99 |
| | $2.0\times10^{-2}$ | | Zhang et al. (2010) | Q | 287, 288 |
| | $6.2\times10^{-3}$ | | Zhang et al. (2010) | Q | 287, 289 |
| | 2.2 | | Zhang et al. (2010) | Q | 287, 290 |
| | $4.2\times10^{-1}$ | | Zhang et al. (2010) | Q | 287, 291 |



Table A6.1: Chlorocarbons (C, H, Cl) (. . . continued)

| Substance Formula (Trivial Name) [CAS Registry Number] InChIKey | $H_s^{cp}$ (at $T^{\ominus}$) $\left[\dfrac{\text{mol}}{\text{m}^3\,\text{Pa}}\right]$ | $\dfrac{\text{d}\ln H_s^{cp}}{\text{d}(1/T)}$ [K] | Reference | Type | Note |
|---|---|---|---|---|---|
| dienochlor $C_{10}Cl_{10}$ [2227-17-0] LWLJUMBEZJHXHV-UHFFFAOYSA-N | $1.8\times10^{-1}$ $1.2\times10^{-2}$ | | Duchowicz et al. (2020) Duchowicz et al. (2020) | V Q | 186 |
| mirex $C_{10}Cl_{12}$ (dodecachloropentacyclodecane) [2385-85-5] GVYLCNUFSHDAAW-UHFFFAOYSA-N | $1.2\times10^{-2}$ $1.2\times10^{-3}$ $5.8\times10^{-2}$ $1.2\times10^{-3}$ $5.2\times10^{-2}$ $9.9\times10^{-4}$ $1.3\times10^{-1}$ 4.3 $3.9\times10^{-3}$ $1.2\times10^{-2}$ | 11000 10000 11000 | Yin and Hassett (1986) Mackay et al. (2006d) McLachlan et al. (1990) Suntio et al. (1988) Smith and Bomberger (1980) Suntio et al. (1988) Keshavarz et al. (2022) Duchowicz et al. (2020) Hilal et al. (2008) Kühne et al. (2005) Duchowicz et al. (2020) Kühne et al. (2005) | M V V V V C Q Q Q Q ? ? | 373 12 24 12 185, 21 |
| dechlorane plus $C_{18}H_{12}Cl_{12}$ [13560-89-9] UGQQAJOWXNCOPY-UHFFFAOYSA-N | 1.3 1.3 $7.0\times10^{-2}$ $2.1\times10^{3}$ $4.6\times10^{1}$ | | HSDB (2015) Zhang et al. (2010) Zhang et al. (2010) Zhang et al. (2010) Zhang et al. (2010) | Q Q Q Q Q | 99 287, 288 287, 289 287, 290 287, 291 |
| chloroethene $CH_2CHCl$ (vinyl chloride) [75-01-4] BZHJMEDXRYGGRV-UHFFFAOYSA-N | $4.1\times10^{-4}$ $3.8\times10^{-4}$ $3.8\times10^{-4}$ $4.3\times10^{-4}$ $3.8\times10^{-4}$ $3.9\times10^{-4}$ $3.9\times10^{-4}$ $4.5\times10^{-4}$ $3.9\times10^{-4}$ $3.9\times10^{-4}$ $4.1\times10^{-4}$ $4.0\times10^{-4}$ $3.7\times10^{-4}$ $8.5\times10^{-6}$ $9.1\times10^{-4}$ $1.2\times10^{-4}$ $1.1\times10^{-4}$ $5.4\times10^{-4}$ $9.4\times10^{-6}$ $4.2\times10^{-4}$ $1.8\times10^{-4}$ $4.4\times10^{-4}$ $6.5\times10^{-4}$ | 2500 3100 3100 3000 3100 3100 3100 3000 2400 3200 2300 2900 3300 | Schwardt et al. (2021) Burkholder et al. (2019) Burkholder et al. (2015) Brockbank (2013) Warneck (2007) Staudinger and Roberts (2001) Staudinger and Roberts (1996) Wilhelm et al. (1977) Schwardt et al. (2021) Hiatt (2013) Chen et al. (2012) Chiang et al. (1998) Ashworth et al. (1988) Gossett (1987) Pearson and McConnell (1975) Mackay et al. (2006b) Lide and Frederikse (1995) Mackay et al. (1993) Hwang et al. (1992) Smith and Bomberger (1980) Dilling (1977) Dilling (1977) Hine and Mookerjee (1975) Yaws (2003) Ryan et al. (1988) | L L L L L L L L M M M M M M M V V V V V V V V X C | 1 1 682, 11 652 278 649, 373 683 24 237 |



Table A6.1: Chlorocarbons (C, H, Cl) (...continued)

| Substance Formula (Trivial Name) [CAS Registry Number] InChIKey | $H_s^{cp}$ (at $T^{\ominus}$) $\left[\dfrac{\mathrm{mol}}{\mathrm{m^3\,Pa}}\right]$ | $\dfrac{\mathrm{d\ln} H_s^{cp}}{\mathrm{d}(1/T)}$ [K] | Reference | Type | Note |
|---|---|---|---|---|---|
| | $6.0\times10^{-4}$ | | Wang et al. (2017) | Q | 80, 238 |
| | $2.4\times10^{-4}$ | | Wang et al. (2017) | Q | 80, 239 |
| | $6.3\times10^{-4}$ | | Wang et al. (2017) | Q | 80, 240 |
| | $3.4\times10^{-3}$ | | Li et al. (2014) | Q | 241 |
| | $4.9\times10^{-4}$ | | Gharagheizi et al. (2012) | Q | |
| | $6.2\times10^{-4}$ | | Raventos-Duran et al. (2010) | Q | 271, 243 |
| | $2.5\times10^{-4}$ | | Raventos-Duran et al. (2010) | Q | 244 |
| | $2.5\times10^{-4}$ | | Raventos-Duran et al. (2010) | Q | 245 |
| | $2.6\times10^{-4}$ | | Gharagheizi et al. (2010) | Q | 246 |
| | $2.1\times10^{-4}$ | | Hilal et al. (2008) | Q | |
| | $1.2\times10^{-3}$ | | Modarresi et al. (2007) | Q | 67 |
| | $4.2\times10^{-4}$ | | Yaffe et al. (2003) | Q | 248, 249 |
| | $1.4\times10^{-3}$ | | Yao et al. (2002) | Q | 229 |
| | $2.4\times10^{-3}$ | | Suzuki et al. (1992) | Q | 232 |
| | $2.0\times10^{-3}$ | | Nirmalakhandan and Speece (1988) | Q | |
| | $8.1\times10^{-4}$ | | Irmann (1965) | Q | |
| | $3.7\times10^{-4}$ | | Mackay et al. (2006b) | ? | |
| | $4.4\times10^{-4}$ | | Yaws (1999) | ? | 21 |
| | $3.9\times10^{-4}$ | 2700 | Yaws et al. (1999) | ? | 21 |
| | $3.7\times10^{-4}$ | | Mackay et al. (1993) | ? | |
| | $4.4\times10^{-4}$ | | Yaws and Yang (1992) | ? | 21 |
| | $4.5\times10^{-4}$ | | Abraham et al. (1990) | ? | |
| chloroethene-d3 CD$_2$CDCl (vinyl chloride-d3) [6745-35-3] BZHJMEDXRYGGRV-FUDHJZNOSA-N | $3.8\times10^{-4}$ | 3100 | Hiatt (2013) | M | |
| 1,1-dichloroethene CH$_2$CCl$_2$ [75-35-4] LGXVIGDEPROXKC-UHFFFAOYSA-N | $3.7\times10^{-4}$ | 3500 | Schwardt et al. (2021) | L | 1 |
| | $3.7\times10^{-4}$ | 3400 | Burkholder et al. (2019) | L | |
| | $3.7\times10^{-4}$ | 3400 | Burkholder et al. (2015) | L | |
| | $3.1\times10^{-4}$ | 3900 | Brockbank (2013) | L | 1, 684 |
| | $3.7\times10^{-4}$ | 3400 | Warneck (2007) | L | |
| | $4.0\times10^{-4}$ | 3800 | Fogg and Sangster (2003) | L | |
| | $3.4\times10^{-4}$ | 4000 | Staudinger and Roberts (2001) | L | |
| | $3.4\times10^{-4}$ | 3900 | Staudinger and Roberts (1996) | L | |
| | $4.0\times10^{-4}$ | 3400 | Schwardt et al. (2021) | M | 685 |
| | $4.1\times10^{-4}$ | 4600 | Hiatt (2013) | M | |
| | $3.7\times10^{-4}$ | 4200 | Dewulf et al. (1999) | M | |
| | $4.4\times10^{-4}$ | | Chiang et al. (1998) | M | 12 |
| | $4.6\times10^{-4}$ | 1600 | Kondoh and Nakajima (1997) | M | |
| | $3.5\times10^{-4}$ | 3300 | Tse et al. (1992) | M | |
| | $3.4\times10^{-4}$ | 4500 | Bissonette et al. (1990) | M | |
| | $3.7\times10^{-4}$ | 2900 | Ashworth et al. (1988) | M | 278, 686 |
| | $3.8\times10^{-4}$ | 3700 | Gossett (1987) | M | |
| | $1.3\times10^{-4}$ | | Yurteri et al. (1987) | M | 12 |
| | $2.6\times10^{-4}$ | 4600 | Leighton and Calo (1981) | M | |





Table A6.1: Chlorocarbons (C, H, Cl) (...continued)

| Substance<br>Formula<br>(Trivial Name)<br>[CAS Registry Number]<br><small>InChIKey</small> | $H_s^{cp}$<br>(at $T^{\ominus}$)<br>$\left[\dfrac{\mathrm{mol}}{\mathrm{m^3\,Pa}}\right]$ | $\dfrac{\mathrm{d}\ln H_s^{cp}}{\mathrm{d}(1/T)}$<br><br>[K] | Reference | Type | Note |
|---|---|---|---|---|---|
| | $1.4\times10^{-4}$ | 6600 | Ervin et al. (1980) | M | |
| | $6.6\times10^{-4}$ | | Warner et al. (1980) | M | |
| | $6.6\times10^{-5}$ | | Pearson and McConnell (1975) | M | 649, 12 |
| | $4.3\times10^{-4}$ | | Mackay et al. (2006b) | V | |
| | $3.3\times10^{-4}$ | | Lide and Frederikse (1995) | V | |
| | $4.3\times10^{-4}$ | | Mackay et al. (1993) | V | |
| | $7.5\times10^{-5}$ | | Mackay and Shiu (1981) | V | |
| | $6.5\times10^{-4}$ | | Warner et al. (1980) | V | |
| | $5.2\times10^{-5}$ | | Dilling (1977) | V | 651 |
| | $6.1\times10^{-5}$ | | Dilling (1977) | V | 12 |
| | $4.3\times10^{-4}$ | | Yaws (2003) | X | 237 |
| | $6.4\times10^{-4}$ | 1200 | Goldstein (1982) | X | 298 |
| | $2.2\times10^{-3}$ | | Ryan et al. (1988) | C | |
| | $6.6\times10^{-4}$ | | Shen (1982) | C | |
| | $1.0\times10^{-3}$ | | Wang et al. (2017) | Q | 80, 238 |
| | $1.3\times10^{-4}$ | | Wang et al. (2017) | Q | 80, 239 |
| | $4.5\times10^{-4}$ | | Wang et al. (2017) | Q | 80, 240 |
| | $1.3\times10^{-3}$ | | Gharagheizi et al. (2012) | Q | |
| | $6.2\times10^{-4}$ | | Raventos-Duran et al. (2010) | Q | 242, 243 |
| | $1.6\times10^{-4}$ | | Raventos-Duran et al. (2010) | Q | 244 |
| | $3.1\times10^{-4}$ | | Raventos-Duran et al. (2010) | Q | 245 |
| | $1.8\times10^{-4}$ | | Gharagheizi et al. (2010) | Q | 246 |
| | $1.3\times10^{-4}$ | | Hilal et al. (2008) | Q | |
| | $8.9\times10^{-4}$ | | Modarresi et al. (2007) | Q | 67 |
| | | 3300 | Kühne et al. (2005) | Q | |
| | $3.8\times10^{-4}$ | | Yaffe et al. (2003) | Q | 248, 249 |
| | $2.6\times10^{-3}$ | | Yao et al. (2002) | Q | 229, 267 |
| | $2.7\times10^{-4}$ | | Katritzky et al. (1998) | Q | |
| | $3.8\times10^{-4}$ | | Mackay et al. (2006b) | ? | |
| | | 3700 | Kühne et al. (2005) | ? | |
| | $4.3\times10^{-4}$ | | Yaws (1999) | ? | 21 |
| | $3.8\times10^{-4}$ | | Mackay et al. (1993) | ? | |
| | $4.3\times10^{-4}$ | | Yaws and Yang (1992) | ? | 21 |
| | $2.7\times10^{-4}$ | | Abraham et al. (1990) | ? | |
| 1,2-dichloroethene<br>$C_2H_2Cl_2$<br>[540-59-0]<br><small>KFUSEUYYWQURPO-UHFFFAOYSA-N</small> | $1.2\times10^{-2}$ | | Keshavarz et al. (2022) | Q | |
| | $5.1\times10^{-3}$ | | Duchowicz et al. (2020) | Q | 299 |
| | $3.7\times10^{-4}$ | | Hilal et al. (2008) | Q | |
| | $2.3\times10^{-3}$ | | Modarresi et al. (2007) | Q | 67 |
| | $1.4\times10^{-3}$ | | Yaffe et al. (2003) | Q | 248, 272 |
| | $6.2\times10^{-4}$ | | Katritzky et al. (1998) | Q | |
| | $4.5\times10^{-3}$ | | Nirmalakhandan and Speece (1988) | Q | |
| | $2.4\times10^{-3}$ | | Duchowicz et al. (2020) | ? | 185, 21 |



Table A6.1: Chlorocarbons (C, H, Cl) (...continued)

| Substance<br>Formula<br>(Trivial Name)<br>[CAS Registry Number]<br>InChIKey | $H_s^{cp}$<br>(at $T^{\ominus}$)<br>$\left[\dfrac{\text{mol}}{\text{m}^3\,\text{Pa}}\right]$ | $\dfrac{\text{d}\ln H_s^{cp}}{\text{d}(1/T)}$<br><br>[K] | Reference | Type | Note |
|---|---|---|---|---|---|
| ($Z$)-1,2-dichloroethene | $2.3\times10^{-3}$ | 3400 | Schwardt et al. (2021) | L | 1 |
| CHClCHCl | $2.6\times10^{-3}$ | 3700 | Burkholder et al. (2019) | L | |
| (*cis*-1,2-dichloroethene) | $2.6\times10^{-3}$ | 3700 | Burkholder et al. (2015) | L | |
| [156-59-2] | $2.4\times10^{-3}$ | 4000 | Brockbank (2013) | L | 1 |
| KFUSEUYYWQURPO-UPHRSURJSA-N | $2.6\times10^{-3}$ | 3700 | Warneck (2007) | L | |
| | $2.5\times10^{-3}$ | 4000 | Fogg and Sangster (2003) | L | |
| | $2.3\times10^{-3}$ | 3900 | Staudinger and Roberts (2001) | L | |
| | $2.3\times10^{-3}$ | 3900 | Staudinger and Roberts (1996) | L | |
| | $2.7\times10^{-3}$ | 3800 | Hiatt (2013) | M | |
| | $2.5\times10^{-3}$ | 3900 | Chen et al. (2012) | M | |
| | $2.2\times10^{-3}$ | 3100 | Shimotori and Arnold (2003) | M | |
| | $1.5\times10^{-3}$ | | Ryu and Park (1999) | M | |
| | $3.2\times10^{-3}$ | | Hovorka and Dohnal (1997) | M | 12 |
| | $2.5\times10^{-3}$ | 3000 | Kondoh and Nakajima (1997) | M | |
| | $1.3\times10^{-3}$ | 3100 | Park et al. (1997) | M | |
| | $2.4\times10^{-3}$ | 4000 | Wright et al. (1992) | M | 687 |
| | $2.5\times10^{-3}$ | 3800 | Tse et al. (1992) | M | |
| | $2.5\times10^{-3}$ | 4200 | Bissonette et al. (1990) | M | |
| | $2.1\times10^{-3}$ | 3200 | Ashworth et al. (1988) | M | 278 |
| | $2.6\times10^{-3}$ | 4200 | Gossett (1987) | M | |
| | $2.2\times10^{-3}$ | | Yurteri et al. (1987) | M | 12 |
| | $2.2\times10^{-3}$ | 4100 | Ervin et al. (1980) | M | |
| | $1.1\times10^{-3}$ | | Sato and Nakajima (1979b) | M | 14 |
| | $1.3\times10^{-3}$ | | Mackay et al. (2006b) | V | |
| | $1.3\times10^{-3}$ | | Park et al. (1997) | V | |
| | $1.3\times10^{-3}$ | | Mackay et al. (1993) | V | |
| | $1.3\times10^{-3}$ | | Mackay and Shiu (1981) | V | |
| | $1.3\times10^{-3}$ | | Dilling (1977) | V | |
| | $2.9\times10^{-3}$ | | Hine and Mookerjee (1975) | V | |
| | $1.3\times10^{-3}$ | | Yaws (2003) | X | 237 |
| | $1.8\times10^{-3}$ | | Wang et al. (2017) | Q | 314, 80, 238 |
| | $4.9\times10^{-4}$ | | Wang et al. (2017) | Q | 314, 80, 239 |
| | $1.1\times10^{-3}$ | | Wang et al. (2017) | Q | 314, 80, 240 |
| | $3.0\times10^{-3}$ | | Gharagheizi et al. (2012) | Q | |
| | $1.4\times10^{-3}$ | | Gharagheizi et al. (2010) | Q | 246 |
| | $2.3\times10^{-3}$ | | Modarresi et al. (2007) | Q | 67 |
| | | 3300 | Kühne et al. (2005) | Q | |
| | $1.4\times10^{-3}$ | | Yaffe et al. (2003) | Q | 248, 249 |
| | $5.2\times10^{-3}$ | | Yao et al. (2002) | Q | 229 |
| | $2.7\times10^{-3}$ | | English and Carroll (2001) | Q | 230, 231 |
| | $2.2\times10^{-3}$ | | Mackay et al. (2006b) | ? | |
| | | 4200 | Kühne et al. (2005) | ? | |
| | $1.3\times10^{-3}$ | | Yaws (1999) | ? | 21 |
| | $1.2\times10^{-3}$ | | Abraham and Weathersby (1994) | ? | 21 |



Table A6.1: Chlorocarbons (C, H, Cl) (...continued)

| Substance Formula (Trivial Name) [CAS Registry Number] InChIKey | $H_s^{cp}$ (at $T^\ominus$) $\left[\dfrac{\mathrm{mol}}{\mathrm{m^3\,Pa}}\right]$ | $\dfrac{\mathrm{d}\ln H_s^{cp}}{\mathrm{d}(1/T)}$ [K] | Reference | Type | Note |
|---|---|---|---|---|---|
| | $2.2\times10^{-3}$ | | Mackay et al. (1993) | ? | |
| | $1.3\times10^{-3}$ | | Yaws and Yang (1992) | ? | 21 |
| | $1.3\times10^{-3}$ | | Abraham et al. (1990) | ? | |
| (*E*)-1,2-dichloroethene | $9.9\times10^{-4}$ | 3600 | Schwardt et al. (2021) | L | 1 |
| CHClCHCl | $1.0\times10^{-3}$ | 3600 | Burkholder et al. (2019) | L | |
| (*trans*-1,2-dichloroethene) | $1.0\times10^{-3}$ | 3600 | Burkholder et al. (2015) | L | |
| [156-60-5] | $1.0\times10^{-3}$ | 3900 | Brockbank (2013) | L | 1 |
| KFUSEUYYWQURPO-OWOJBTEDSA-N | $1.0\times10^{-3}$ | 3500 | Warneck (2007) | L | |
| | $1.1\times10^{-3}$ | 4200 | Fogg and Sangster (2003) | L | |
| | $9.0\times10^{-4}$ | 4100 | Staudinger and Roberts (2001) | L | |
| | $9.0\times10^{-4}$ | 4100 | Staudinger and Roberts (1996) | L | |
| | $9.2\times10^{-4}$ | 2600 | Schwardt et al. (2021) | M | 688, 11 |
| | $1.0\times10^{-3}$ | 4000 | Hiatt (2013) | M | |
| | $1.0\times10^{-3}$ | 3500 | Shimotori and Arnold (2003) | M | |
| | $1.6\times10^{-3}$ | | Ryu and Park (1999) | M | |
| | $1.3\times10^{-3}$ | | Hovorka and Dohnal (1997) | M | 12 |
| | $1.1\times10^{-3}$ | 2200 | Kondoh and Nakajima (1997) | M | |
| | $1.8\times10^{-3}$ | 6200 | Park et al. (1997) | M | |
| | $9.3\times10^{-4}$ | 4900 | Khalfaoui and Newsham (1994b) | M | 689 |
| | $9.8\times10^{-4}$ | 3400 | Hansen et al. (1993) | M | 281 |
| | $1.0\times10^{-3}$ | 4200 | Wright et al. (1992) | M | 690 |
| | $1.0\times10^{-3}$ | 3700 | Tse et al. (1992) | M | |
| | $9.7\times10^{-4}$ | 5000 | Cooling et al. (1992) | M | 691 |
| | $8.4\times10^{-4}$ | 4800 | Bissonette et al. (1990) | M | |
| | $9.9\times10^{-4}$ | 3000 | Ashworth et al. (1988) | M | 278, 686 |
| | $1.1\times10^{-3}$ | 4200 | Gossett (1987) | M | |
| | $1.1\times10^{-3}$ | | Yurteri et al. (1987) | M | 12 |
| | $7.0\times10^{-4}$ | 5400 | Ervin et al. (1980) | M | |
| | $1.9\times10^{-3}$ | | Warner et al. (1980) | M | |
| | $8.1\times10^{-4}$ | | Sato and Nakajima (1979b) | M | 14 |
| | $1.5\times10^{-3}$ | | Mackay et al. (2006b) | V | |
| | $1.5\times10^{-3}$ | | Park et al. (1997) | V | |
| | $1.5\times10^{-3}$ | | Mackay et al. (1993) | V | |
| | $1.5\times10^{-3}$ | | Hwang et al. (1992) | V | |
| | $1.5\times10^{-3}$ | | Mackay and Shiu (1981) | V | |
| | $2.4\times10^{-3}$ | | Warner et al. (1980) | V | |
| | $1.5\times10^{-3}$ | | Dilling (1977) | V | |
| | $1.5\times10^{-3}$ | | Hine and Mookerjee (1975) | V | |
| | $1.5\times10^{-3}$ | | Yaws (2003) | X | 237 |
| | $1.9\times10^{-3}$ | 1700 | Goldstein (1982) | X | 298 |
| | $1.5\times10^{-3}$ | | Ryan et al. (1988) | C | |
| | $1.9\times10^{-3}$ | | Shen (1982) | C | |
| | $1.8\times10^{-3}$ | | Wang et al. (2017) | Q | 314, 80, 238 |
| | $4.9\times10^{-4}$ | | Wang et al. (2017) | Q | 314, 80, 239 |



Table A6.1: Chlorocarbons (C, H, Cl) (...continued)

| Substance<br>Formula<br>(Trivial Name)<br>[CAS Registry Number]<br>InChIKey | $H_s^{cp}$<br>(at $T^\ominus$)<br>$\left[\dfrac{\mathrm{mol}}{\mathrm{m^3\,Pa}}\right]$ | $\dfrac{\mathrm{d}\ln H_s^{cp}}{\mathrm{d}(1/T)}$<br><br>[K] | Reference | Type | Note |
|---|---|---|---|---|---|
| | $1.1\times10^{-3}$ | | Wang et al. (2017) | Q | 314, 80, 240 |
| | $2.1\times10^{-3}$ | | Gharagheizi et al. (2012) | Q | |
| | $1.4\times10^{-3}$ | | Gharagheizi et al. (2010) | Q | 246 |
| | $2.3\times10^{-3}$ | | Modarresi et al. (2007) | Q | 67 |
| | | 3300 | Kühne et al. (2005) | Q | |
| | $2.2\times10^{-3}$ | | Yao et al. (2002) | Q | 229 |
| | $2.3\times10^{-3}$ | | English and Carroll (2001) | Q | 230, 231 |
| | $1.0\times10^{-3}$ | | Mackay et al. (2006b) | ? | |
| | | 4300 | Kühne et al. (2005) | ? | |
| | $1.5\times10^{-3}$ | | Yaws (1999) | ? | 21 |
| | $8.4\times10^{-4}$ | | Abraham and Weathersby (1994) | ? | 21 |
| | $1.0\times10^{-3}$ | | Mackay et al. (1993) | ? | |
| | $1.5\times10^{-3}$ | | Yaws and Yang (1992) | ? | 21 |
| | $1.5\times10^{-3}$ | | Abraham et al. (1990) | ? | |
| trichloroethene | $1.1\times10^{-3}$ | 4100 | Schwardt et al. (2021) | L | 1 |
| C$_2$HCl$_3$ | $1.1\times10^{-3}$ | 4300 | Burkholder et al. (2019) | L | |
| (trichloroethylene) | $8.6\times10^{-4}$ | 4200 | Burkholder et al. (2019) | L | 70 |
| [79-01-6] | $1.1\times10^{-3}$ | 4300 | Burkholder et al. (2015) | L | |
| XSTXAVWGXDQKEL-UHFFFAOYSA-N | $8.6\times10^{-4}$ | 4200 | Burkholder et al. (2015) | L | 70 |
| | $1.0\times10^{-3}$ | 4200 | Brockbank (2013) | L | 1 |
| | $1.1\times10^{-3}$ | 4300 | Warneck (2007) | L | |
| | $1.0\times10^{-3}$ | 4300 | Fogg and Sangster (2003) | L | |
| | $1.0\times10^{-3}$ | 4600 | Staudinger and Roberts (2001) | L | |
| | $9.9\times10^{-4}$ | 4600 | Staudinger and Roberts (1996) | L | |
| | $6.6\times10^{-4}$ | | Steward et al. (1973) | L | 14 |
| | $1.1\times10^{-3}$ | 4100 | Allott et al. (1973) | L | |
| | $1.0\times10^{-3}$ | 4200 | Schwardt et al. (2021) | M | 692 |
| | $1.2\times10^{-3}$ | 4700 | Hiatt (2013) | M | |
| | $1.6\times10^{-3}$ | 2800 | Zhang et al. (2013) | M | 324 |
| | $1.3\times10^{-3}$ | | Zhang et al. (2013) | M | 325 |
| | $1.0\times10^{-3}$ | 3900 | Chen et al. (2012) | M | |
| | $9.4\times10^{-4}$ | | Helburn et al. (2008) | M | |
| | $1.0\times10^{-3}$ | 3900 | Shimotori and Arnold (2003) | M | |
| | $9.5\times10^{-4}$ | 4300 | Görgényi et al. (2002) | M | 693 |
| | $1.2\times10^{-3}$ | 3600 | Bierwagen and Keller (2001) | M | |
| | $7.6\times10^{-4}$ | 4900 | Moore (2000) | M | 70 |
| | $1.0\times10^{-3}$ | | David et al. (2000) | M | 72 |
| | $1.1\times10^{-3}$ | 3900 | Vane and Giroux (2000) | M | |
| | $1.1\times10^{-3}$ | 4800 | Knauss et al. (2000) | M | 694 |
| | $9.5\times10^{-4}$ | 4900 | Dewulf et al. (1999) | M | |
| | $9.5\times10^{-4}$ | | Ryu and Park (1999) | M | |
| | $9.3\times10^{-4}$ | 3700 | Heron et al. (1998) | M | |
| | $1.1\times10^{-3}$ | | Chiang et al. (1998) | M | 12 |
| | $1.4\times10^{-3}$ | | Peng and Wan (1998) | M | |
| | $8.7\times10^{-4}$ | 4000 | Peng and Wan (1998) | M | 70 |
| | $1.1\times10^{-3}$ | 3800 | Peng and Wan (1997) | M | |



Table A6.1: Chlorocarbons (C, H, Cl) (...continued)

| Substance Formula (Trivial Name) [CAS Registry Number] InChIKey | $H_s^{cp}$ (at $T^{\ominus}$) $\left[\dfrac{\mathrm{mol}}{\mathrm{m^3\,Pa}}\right]$ | $\dfrac{\mathrm{d}\ln H_s^{cp}}{\mathrm{d}(1/T)}$ [K] | Reference | Type | Note |
|---|---|---|---|---|---|
| | $1.3\times10^{-3}$ | | Hovorka and Dohnal (1997) | M | 12 |
| | $1.1\times10^{-3}$ | 2200 | Kondoh and Nakajima (1997) | M | |
| | $8.8\times10^{-4}$ | 3600 | Park et al. (1997) | M | |
| | $8.5\times10^{-4}$ | | Turner et al. (1996) | M | |
| | $8.3\times10^{-4}$ | | Ramachandran et al. (1996) | M | |
| | $1.2\times10^{-3}$ | 3900 | Dewulf et al. (1995) | M | |
| | $1.3\times10^{-3}$ | | Nielsen et al. (1994) | M | |
| | $9.5\times10^{-4}$ | 5000 | Khalfaoui and Newsham (1994b) | M | 695 |
| | $9.4\times10^{-4}$ | 3100 | Robbins et al. (1993) | M | 696 |
| | $1.1\times10^{-3}$ | | Hoff et al. (1993) | M | |
| | $1.0\times10^{-3}$ | | Li et al. (1993) | M | |
| | $1.1\times10^{-3}$ | 3700 | Wright et al. (1992) | M | 697 |
| | $1.1\times10^{-3}$ | 4200 | Tse et al. (1992) | M | |
| | $9.7\times10^{-4}$ | 4900 | Cooling et al. (1992) | M | 698 |
| | $1.3\times10^{-3}$ | 5200 | Tancrède and Yanagisawa (1990) | M | |
| | $1.0\times10^{-3}$ | 5200 | Bissonette et al. (1990) | M | |
| | $9.7\times10^{-4}$ | 2000 | Lamarche and Droste (1989) | M | 345 |
| | $5.5\times10^{-4}$ | | Guitart et al. (1989) | M | 14 |
| | $9.5\times10^{-4}$ | 3700 | Ashworth et al. (1988) | M | 278 |
| | $1.0\times10^{-3}$ | 4800 | Gossett (1987) | M | |
| | $9.6\times10^{-4}$ | 4700 | Munz and Roberts (1987) | M | |
| | $9.8\times10^{-4}$ | | Hellmann (1987) | M | 87 |
| | $9.4\times10^{-4}$ | | Yurteri et al. (1987) | M | 12 |
| | $9.0\times10^{-4}$ | 5400 | Schoene and Steinhanses (1985) | M | |
| | $1.1\times10^{-3}$ | 4300 | Gossett et al. (1985) | M | |
| | $1.0\times10^{-3}$ | | Garbarini and Lion (1985) | M | |
| | $9.7\times10^{-4}$ | 4900 | Lincoff and Gossett (1984) | M | |
| | $1.0\times10^{-3}$ | 4600 | Leighton and Calo (1981) | M | |
| | $7.4\times10^{-4}$ | 4800 | Ervin et al. (1980) | M | |
| | $8.4\times10^{-4}$ | | Warner et al. (1980) | M | |
| | $5.0\times10^{-4}$ | | Sato and Nakajima (1979b) | M | 14 |
| | $1.1\times10^{-3}$ | | Pearson and McConnell (1975) | M | 649, 12 |
| | $8.5\times10^{-4}$ | | Mackay et al. (2006b) | V | |
| | $9.9\times10^{-4}$ | | Park et al. (1997) | V | |
| | $8.4\times10^{-4}$ | | Mackay et al. (1993) | V | |
| | $1.1\times10^{-3}$ | | Hwang et al. (1992) | V | |
| | $8.1\times10^{-4}$ | | Mackay and Shiu (1981) | V | |
| | $8.4\times10^{-4}$ | | Warner et al. (1980) | V | |
| | $8.2\times10^{-4}$ | | Dilling (1977) | V | 651 |
| | $1.0\times10^{-3}$ | | Dilling (1977) | V | 12 |
| | $2.4\times10^{-3}$ | | Dilling (1977) | V | 153 |
| | $8.4\times10^{-4}$ | | Hine and Mookerjee (1975) | V | |
| | $8.4\times10^{-4}$ | | Dilling et al. (1975) | V | |
| | $8.6\times10^{-4}$ | | Yaws (2003) | X | 258 |
| | $8.5\times10^{-4}$ | | Yaws (2003) | X | 237 |
| | $8.8\times10^{-4}$ | 1600 | Goldstein (1982) | X | 298 |
| | $1.1\times10^{-3}$ | | Ryan et al. (1988) | C | |





Table A6.1: Chlorocarbons (C, H, Cl) (...continued)

| Substance Formula (Trivial Name) [CAS Registry Number] InChIKey | $H_s^{cp}$ (at $T^\ominus$) $\left[\dfrac{\text{mol}}{\text{m}^3\,\text{Pa}}\right]$ | $\dfrac{\text{d}\ln H_s^{cp}}{\text{d}(1/T)}$ [K] | Reference | Type | Note |
|---|---|---|---|---|---|
| | $8.4\times10^{-4}$ | | Shen (1982) | C | |
| | $6.2\times10^{-4}$ | | Dupeux et al. (2022) | Q | 259 |
| | $2.2\times10^{-3}$ | | Keshavarz et al. (2022) | Q | |
| | $2.9\times10^{-3}$ | | Duchowicz et al. (2020) | Q | 184 |
| | $2.9\times10^{-3}$ | | Wang et al. (2017) | Q | 80, 238 |
| | $2.2\times10^{-4}$ | | Wang et al. (2017) | Q | 80, 239 |
| | $6.9\times10^{-4}$ | | Wang et al. (2017) | Q | 80, 240 |
| | $8.4\times10^{-4}$ | | Li et al. (2014) | Q | 241 |
| | $5.5\times10^{-3}$ | | Gharagheizi et al. (2012) | Q | |
| | $2.5\times10^{-4}$ | | Raventos-Duran et al. (2010) | Q | 271, 243 |
| | $2.0\times10^{-4}$ | | Raventos-Duran et al. (2010) | Q | 244 |
| | $3.9\times10^{-4}$ | | Raventos-Duran et al. (2010) | Q | 245 |
| | $8.7\times10^{-4}$ | | Gharagheizi et al. (2010) | Q | 246 |
| | $3.0\times10^{-4}$ | | Hilal et al. (2008) | Q | |
| | $1.8\times10^{-3}$ | | Modarresi et al. (2007) | Q | 67 |
| | | 3600 | Kühne et al. (2005) | Q | |
| | $1.1\times10^{-3}$ | | Yaffe et al. (2003) | Q | 248, 249 |
| | $1.1\times10^{-3}$ | | English and Carroll (2001) | Q | 230, 274 |
| | $4.0\times10^{-4}$ | | Katritzky et al. (1998) | Q | |
| | $8.4\times10^{-3}$ | | Nirmalakhandan and Speece (1988) | Q | |
| | $1.0\times10^{-3}$ | | Duchowicz et al. (2020) | ? | 185, 21 |
| | $9.7\times10^{-4}$ | | Mackay et al. (2006b) | ? | |
| | | 4200 | Kühne et al. (2005) | ? | |
| | $8.5\times10^{-4}$ | | Yaws (1999) | ? | 21 |
| | $5.2\times10^{-4}$ | | Abraham and Weathersby (1994) | ? | 21 |
| | $9.7\times10^{-4}$ | | Mackay et al. (1993) | ? | |
| | $8.4\times10^{-4}$ | | Yaws and Yang (1992) | ? | 21 |
| | $8.4\times10^{-4}$ | | Abraham et al. (1990) | ? | |
| tetrachloroethene $C_2Cl_4$ (tetrachloroethylene) [127-18-4] CYTYCFOTNPOANT-UHFFFAOYSA-N | $5.7\times10^{-4}$ | 4700 | Schwardt et al. (2021) | L | 1 |
| | $5.5\times10^{-4}$ | 4500 | Burkholder et al. (2019) | L | |
| | $4.6\times10^{-4}$ | 4400 | Burkholder et al. (2019) | L | 70 |
| | $5.5\times10^{-4}$ | 4500 | Burkholder et al. (2015) | L | |
| | $4.6\times10^{-4}$ | 4400 | Burkholder et al. (2015) | L | 70 |
| | $5.6\times10^{-4}$ | 4700 | Brockbank (2013) | L | 1 |
| | $6.2\times10^{-4}$ | 4500 | Warneck (2007) | L | |
| | $6.0\times10^{-4}$ | 4200 | Fogg and Sangster (2003) | L | |
| | $5.9\times10^{-4}$ | 4800 | Staudinger and Roberts (2001) | L | |
| | $5.8\times10^{-4}$ | 4800 | Staudinger and Roberts (1996) | L | |
| | $4.3\times10^{-4}$ | | Mackay and Shiu (1981) | L | |
| | $5.8\times10^{-4}$ | 4500 | Schwardt et al. (2021) | M | 699 |
| | $9.9\times10^{-4}$ | 4600 | Hiatt (2013) | M | |
| | $6.2\times10^{-4}$ | 4200 | Chen et al. (2012) | M | |
| | $5.8\times10^{-4}$ | 4200 | Shimotori and Arnold (2003) | M | |
| | $4.1\times10^{-4}$ | 5300 | Moore (2000) | M | 70 |
| | $6.0\times10^{-4}$ | 4100 | Vane and Giroux (2000) | M | |
| | $4.8\times10^{-4}$ | 4400 | Knauss et al. (2000) | M | 700 |
| | $5.3\times10^{-4}$ | | Ryu and Park (1999) | M | |



Table A6.1: Chlorocarbons (C, H, Cl) (...continued)

| Substance<br>Formula<br>(Trivial Name)<br>[CAS Registry Number]<br>InChIKey | $H_s^{cp}$<br>(at $T^{\ominus}$)<br>$\left[\dfrac{\mathrm{mol}}{\mathrm{m}^3\,\mathrm{Pa}}\right]$ | $\dfrac{\mathrm{d}\ln H_s^{cp}}{\mathrm{d}(1/T)}$<br><br>[K] | Reference | Type | Note |
|---|---|---|---|---|---|
| | $8.6\times10^{-4}$ | | Dohnal and Hovorka (1999) | M | 12 |
| | $6.2\times10^{-4}$ | | Chiang et al. (1998) | M | 12 |
| | $7.8\times10^{-4}$ | | Peng and Wan (1998) | M | |
| | $4.7\times10^{-4}$ | 4100 | Peng and Wan (1998) | M | 70 |
| | $6.1\times10^{-4}$ | 4200 | Peng and Wan (1997) | M | |
| | $8.4\times10^{-4}$ | | Hovorka and Dohnal (1997) | M | 12 |
| | $6.9\times10^{-4}$ | 2200 | Kondoh and Nakajima (1997) | M | |
| | $5.5\times10^{-4}$ | 4200 | Park et al. (1997) | M | |
| | $6.9\times10^{-4}$ | 4800 | Dewulf et al. (1995) | M | |
| | $5.8\times10^{-4}$ | 5200 | Robbins et al. (1993) | M | 701 |
| | $6.3\times10^{-4}$ | | Hoff et al. (1993) | M | |
| | $6.3\times10^{-4}$ | | Li et al. (1993) | M | |
| | $8.1\times10^{-4}$ | 2100 | Kolb et al. (1992) | M | 33, 277 |
| | $5.9\times10^{-4}$ | 5500 | Tancrède and Yanagisawa (1990) | M | |
| | $6.2\times10^{-4}$ | 5300 | Bissonette et al. (1990) | M | |
| | $5.4\times10^{-4}$ | 4400 | Ashworth et al. (1988) | M | 278 |
| | $5.6\times10^{-4}$ | 4900 | Gossett (1987) | M | |
| | $5.4\times10^{-4}$ | 4400 | Munz and Roberts (1987) | M | |
| | $7.7\times10^{-4}$ | | Hellmann (1987) | M | 87 |
| | $7.5\times10^{-4}$ | | Yurteri et al. (1987) | M | 12 |
| | $6.5\times10^{-4}$ | 4600 | Gossett et al. (1985) | M | |
| | $5.7\times10^{-4}$ | 5100 | Lincoff and Gossett (1984) | M | |
| | $6.1\times10^{-4}$ | 4700 | Leighton and Calo (1981) | M | |
| | $5.7\times10^{-4}$ | 5200 | Ervin et al. (1980) | M | |
| | $3.4\times10^{-4}$ | | Warner et al. (1980) | M | |
| | $1.1\times10^{-3}$ | 4300 | Gossett (1980) | M | |
| | $1.7\times10^{-4}$ | | Sato and Nakajima (1979b) | M | 14 |
| | $5.0\times10^{-4}$ | | Pearson and McConnell (1975) | M | 649, 12 |
| | $3.7\times10^{-4}$ | | Mackay et al. (2006b) | V | |
| | $3.4\times10^{-4}$ | | Park et al. (1997) | V | |
| | $3.7\times10^{-4}$ | | Mackay et al. (1993) | V | |
| | $3.6\times10^{-4}$ | | Hwang et al. (1992) | V | |
| | $9.1\times10^{-4}$ | | Addison et al. (1983) | V | |
| | $3.5\times10^{-4}$ | | Warner et al. (1980) | V | |
| | $3.4\times10^{-4}$ | | Dilling (1977) | V | 651 |
| | $4.0\times10^{-4}$ | | Dilling (1977) | V | 12 |
| | $1.2\times10^{-3}$ | | Dilling (1977) | V | 153 |
| | $3.7\times10^{-4}$ | | Hine and Mookerjee (1975) | V | |
| | $9.8\times10^{-4}$ | | Dilling et al. (1975) | V | |
| | $3.6\times10^{-4}$ | | Yaws (2003) | X | 237 |
| | $3.6\times10^{-4}$ | 1500 | Goldstein (1982) | X | 298 |
| | $6.3\times10^{-4}$ | | Ryan et al. (1988) | C | |
| | $3.4\times10^{-4}$ | | Shen (1982) | C | |
| | $8.1\times10^{-4}$ | | Dilling (1977) | C | |
| | $8.1\times10^{-4}$ | | Dilling et al. (1975) | C | |
| | $4.4\times10^{-3}$ | | Wang et al. (2017) | Q | 80, 238 |
| | $1.2\times10^{-4}$ | | Wang et al. (2017) | Q | 80, 239 |



Table A6.1: Chlorocarbons (C, H, Cl) (... continued)

| Substance / Formula / (Trivial Name) / [CAS Registry Number] / InChIKey | $H_s^{cp}$ (at $T^\ominus$) $\left[\dfrac{\text{mol}}{\text{m}^3\,\text{Pa}}\right]$ | $\dfrac{\text{d}\ln H_s^{cp}}{\text{d}(1/T)}$ [K] | Reference | Type | Note |
|---|---|---|---|---|---|
| | $2.5\times10^{-4}$ | | Wang et al. (2017) | Q | 80, 240 |
| | $1.1\times10^{-2}$ | | Gharagheizi et al. (2012) | Q | |
| | $3.9\times10^{-4}$ | | Gharagheizi et al. (2010) | Q | 246 |
| | $1.7\times10^{-4}$ | | Hilal et al. (2008) | Q | |
| | $1.2\times10^{-3}$ | | Modarresi et al. (2007) | Q | 67 |
| | | 3900 | Kühne et al. (2005) | Q | |
| | $5.6\times10^{-4}$ | | Yaffe et al. (2003) | Q | 248, 249 |
| | $3.1\times10^{-4}$ | | English and Carroll (2001) | Q | 230, 231 |
| | $9.9\times10^{-5}$ | | Katritzky et al. (1998) | Q | |
| | $8.8\times10^{-4}$ | | Nirmalakhandan and Speece (1988) | Q | |
| | $5.8\times10^{-4}$ | | Mackay et al. (2006b) | ? | |
| | | 5100 | Kühne et al. (2005) | ? | |
| | $3.7\times10^{-4}$ | | Yaws (1999) | ? | 21 |
| | $1.7\times10^{-4}$ | | Abraham and Weathersby (1994) | ? | 21 |
| | $5.8\times10^{-4}$ | | Mackay et al. (1993) | ? | |
| | $3.7\times10^{-4}$ | | Yaws and Yang (1992) | ? | 21 |
| | $3.4\times10^{-4}$ | | Abraham et al. (1990) | ? | |
| | $2.9\times10^{-3}$ | | Chiou et al. (1980) | ? | 79 |
| dichloroethyne $C_2Cl_2$ [7572-29-4] ZMJOVJSTYLQINE-UHFFFAOYSA-N | $4.9\times10^{-4}$ | | HSDB (2015) | Q | 99 |
| 1-chloro-1-propene $C_3H_5Cl$ [590-21-6] OWXJKYNZGFSVRC-UHFFFAOYSA-N | $1.8\times10^{-4}$ | | HSDB (2015) | Q | 99 |
| 2-chloro-1-propene $C_3H_5Cl$ [557-98-2] PNLQPWWBHXMFCA-UHFFFAOYSA-N | $1.4\times10^{-4}$ | | HSDB (2015) | Q | 99 |
| 3-chloro-1-propene $C_3H_5Cl$ (allyl chloride) [107-05-1] OSDWBNJEKMUWAV-UHFFFAOYSA-N | $9.1\times10^{-4}$ | | Mackay and Shiu (1981) | L | |
| | $1.3\times10^{-3}$ | 4500 | Hiatt (2013) | M | |
| | $3.7\times10^{-3}$ | | Welke et al. (1998) | M | |
| | $9.0\times10^{-4}$ | | Duchowicz et al. (2020) | V | 186 |
| | $9.0\times10^{-4}$ | | HSDB (2015) | V | |
| | $4.6\times10^{-4}$ | | Mackay et al. (1993) | V | |
| | $9.2\times10^{-4}$ | | Dilling (1977) | V | |
| | $1.1\times10^{-3}$ | | Hine and Mookerjee (1975) | V | |
| | $1.1\times10^{-3}$ | | Yaws (2003) | X | 237 |
| | $4.6\times10^{-3}$ | | Duchowicz et al. (2020) | Q | |
| | $1.1\times10^{-3}$ | | Li et al. (2014) | Q | 241 |
| | $1.4\times10^{-3}$ | | Gharagheizi et al. (2012) | Q | |
| | $8.4\times10^{-4}$ | | Gharagheizi et al. (2010) | Q | 246 |
| | $4.0\times10^{-3}$ | | Hilal et al. (2008) | Q | |
| | $2.8\times10^{-3}$ | | Modarresi et al. (2007) | Q | 67 |





Table A6.1: Chlorocarbons (C, H, Cl) (...continued)

| Substance Formula (Trivial Name) [CAS Registry Number] InChIKey | $H_s^{cp}$ (at $T^{\ominus}$) $\left[\dfrac{\text{mol}}{\text{m}^3\,\text{Pa}}\right]$ | $\dfrac{\text{d}\ln H_s^{cp}}{\text{d}(1/T)}$ [K] | Reference | Type | Note |
|---|---|---|---|---|---|
| | $5.9\times10^{-4}$ | | Yao et al. (2002) | Q | 229 |
| | $1.1\times10^{-3}$ | | English and Carroll (2001) | Q | 230, 231 |
| | $1.8\times10^{-3}$ | | Nirmalakhandan et al. (1997) | Q | |
| | $1.9\times10^{-3}$ | | Suzuki et al. (1992) | Q | 232 |
| | $1.7\times10^{-3}$ | | Nirmalakhandan and Speece (1988) | Q | |
| | $1.1\times10^{-3}$ | | Yaws (1999) | ? | 21 |
| | $1.1\times10^{-3}$ | | Yaws and Yang (1992) | ? | 21 |
| | $1.1\times10^{-3}$ | | Abraham et al. (1990) | ? | |
| 1,1-dichloropropene $C_3H_4Cl_2$ [563-58-6] ZAIDIVBQUMFXEC-UHFFFAOYSA-N | $6.1\times10^{-4}$ $5.4\times10^{-4}$ | 4200 1900 | Hiatt (2013) Kondoh and Nakajima (1997) | M M | |
| 1,2-dichloropropene $C_3H_4Cl_2$ [563-54-2] PPKPKFIWDXDAGC-UHFFFAOYSA-N | $2.0\times10^{-3}$ $3.1\times10^{-4}$ $1.8\times10^{-3}$ | | HSDB (2015) Hilal et al. (2008) Modarresi et al. (2007) | V Q Q | 67 |
| 1,3-dichloropropene $C_3H_4Cl_2$ [542-75-6] UOORRWUZONOOLO-UHFFFAOYSA-N | $6.4\times10^{-3}$ $2.8\times10^{-3}$ $7.3\times10^{-3}$ $2.4\times10^{-5}$ $2.8\times10^{-3}$ $5.8\times10^{-3}$ $2.8\times10^{-3}$ $8.1\times10^{-3}$ $2.8\times10^{-3}$ $3.9\times10^{-3}$ $6.2\times10^{-3}$ $3.9\times10^{-4}$ $5.1\times10^{-5}$ $5.7\times10^{-3}$ $5.1\times10^{-3}$ | 4300 1500 | Wright et al. (1992) Warner et al. (1980) Warner et al. (1980) Barcelo and Hennion (1997) Goldstein (1982) Hilal et al. (2008) Horvath and Getzen (1999) Ryan et al. (1988) Shen (1982) Raventos-Duran et al. (2010) Raventos-Duran et al. (2010) Raventos-Duran et al. (2010) Goodarzi et al. (2010) Hilal et al. (2008) Modarresi et al. (2007) | M M V X X C C C C Q Q Q Q Q Q | 702 567 298 242, 243 244 245 568 67 |
| *cis*-1,3-dichloropropene $C_3H_4Cl_2$ [10061-01-5] UOORRWUZONOOLO-UPHRSURJSA-N | $4.2\times10^{-3}$ $9.5\times10^{-3}$ $1.0\times10^{-2}$ $6.3\times10^{-3}$ $4.2\times10^{-3}$ $5.0\times10^{-3}$ $4.2\times10^{-3}$ $8.7\times10^{-3}$ $5.5\times10^{-3}$ | 5500 4300 5800 | Mackay and Shiu (1981) Hiatt (2013) Thomas et al. (2006) Kondoh and Nakajima (1997) Albanese et al. (1987) Leistra (1970) Dilling (1977) Thomas et al. (2006) Yates and Gan (1998) | L M M M M M V ? ? | 154, 703 154, 704 |



Table A6.1: Chlorocarbons (C, H, Cl) (. . . continued)

| Substance<br>Formula<br>(Trivial Name)<br>[CAS Registry Number]<br>InChIKey | $H_s^{cp}$<br>(at $T^{\ominus}$)<br>$\left[\dfrac{\text{mol}}{\text{m}^3\,\text{Pa}}\right]$ | $\dfrac{\text{d}\ln H_s^{cp}}{\text{d}(1/T)}$<br><br>[K] | Reference | Type | Note |
|---|---|---|---|---|---|
| *trans*-1,3-dichloropropene<br>$C_3H_4Cl_2$<br>[10061-02-6]<br>UOORRWUZONOOLO-OWOJBTEDSA-N | $5.6\times10^{-3}$<br>$5.8\times10^{-3}$<br>$1.5\times10^{-2}$<br>$1.0\times10^{-2}$<br>$5.6\times10^{-3}$<br>$8.1\times10^{-3}$<br>$5.6\times10^{-3}$<br>$1.5\times10^{-2}$<br>$9.4\times10^{-3}$ | <br>4800<br><br>5000<br><br>5700 | Mackay and Shiu (1981)<br>Hiatt (2013)<br>Thomas et al. (2006)<br>Kondoh and Nakajima (1997)<br>Albanese et al. (1987)<br>Leistra (1970)<br>Dilling (1977)<br>Thomas et al. (2006)<br>Yates and Gan (1998) | L<br>M<br>M<br>M<br>M<br>M<br>V<br>?<br>? | <br><br>154, 703<br><br><br><br><br>154, 704<br> |
| 2,3-dichloropropene<br>$C_3H_4Cl_2$<br>[78-88-6]<br>FALCMQXTWHPRIH-UHFFFAOYSA-N | $2.8\times10^{-3}$<br>$3.5\times10^{-3}$<br>$2.4\times10^{-3}$<br>$2.7\times10^{-3}$<br>$2.4\times10^{-3}$<br>$4.0\times10^{-3}$<br>$4.6\times10^{-3}$<br>$3.9\times10^{-3}$<br>$6.2\times10^{-3}$<br>$3.9\times10^{-4}$<br>$2.4\times10^{-3}$<br>$4.8\times10^{-3}$<br>$3.8\times10^{-3}$<br>$2.9\times10^{-3}$<br>$1.8\times10^{-3}$<br>$2.0\times10^{-3}$<br>$2.4\times10^{-3}$ | | Mackay and Shiu (1981)<br>Albanese et al. (1987)<br>Duchowicz et al. (2020)<br>Dilling (1977)<br>Yaws (2003)<br>Duchowicz et al. (2020)<br>Gharagheizi et al. (2012)<br>Raventos-Duran et al. (2010)<br>Raventos-Duran et al. (2010)<br>Raventos-Duran et al. (2010)<br>Gharagheizi et al. (2010)<br>Hilal et al. (2008)<br>Modarresi et al. (2007)<br>Yaffe et al. (2003)<br>Yao et al. (2002)<br>Katritzky et al. (1998)<br>Yaws (1999) | L<br>M<br>V<br>V<br>X<br>Q<br>Q<br>Q<br>Q<br>Q<br>Q<br>Q<br>Q<br>Q<br>Q<br>Q<br>? | <br><br>186<br><br>237<br><br><br>242, 243<br>244<br>245<br>246<br><br>67<br>248, 249<br>229, 267<br><br>21 |
| 1,2,3-trichloro-1-propene<br>$C_3H_3Cl_3$<br>[96-19-5]<br>HIILBTHBHCLUER-UHFFFAOYSA-N | $5.5\times10^{-4}$ | | HSDB (2015) | Q | 99 |
| 1,1,2,3,3,3-hexachloro-1-propene<br>$C_3Cl_6$<br>[1888-71-7]<br>VFDYKPARTDCDCU-UHFFFAOYSA-N | $2.1\times10^{-3}$<br>$1.3\times10^{-3}$<br>$6.2\times10^{-3}$<br>$9.9\times10^{-4}$<br>$3.8\times10^{-3}$<br>$2.1\times10^{-3}$ | | Duchowicz et al. (2020)<br>Duchowicz et al. (2020)<br>HSDB (2015)<br>Hilal et al. (2008)<br>Modarresi et al. (2007)<br>Yaffe et al. (2003) | V<br>Q<br>Q<br>Q<br>Q<br>Q | 186<br><br>99<br><br>67<br>248, 249 |
| 3-chloro-1-butene<br>$C_4H_7Cl$<br>[563-52-0]<br>VZGLVCFVUREVDP-UHFFFAOYSA-N | $5.6\times10^{-4}$ | | Ebert et al. (2023) | ? | 316 |
| 3-chloro-2-methyl-1-propene<br>$C_4H_7Cl$<br>[563-47-3]<br>OHXAOPZTJOUYKM-UHFFFAOYSA-N | $1.1\times10^{-3}$<br>$1.1\times10^{-3}$<br>$1.5\times10^{-3}$ | | Duchowicz et al. (2020)<br>HSDB (2015)<br>Duchowicz et al. (2020) | V<br>V<br>Q | 186<br><br> |



Table A6.1: Chlorocarbons (C, H, Cl) (. . . continued)

| Substance Formula (Trivial Name) [CAS Registry Number] InChIKey | $H_s^{cp}$ (at $T^{\ominus}$) $\left[\dfrac{\mathrm{mol}}{\mathrm{m^3\,Pa}}\right]$ | $\dfrac{\mathrm{d}\ln H_s^{cp}}{\mathrm{d}(1/T)}$ [K] | Reference | Type | Note |
|---|---|---|---|---|---|
| 1-chloro-2-butene $C_4H_7Cl$ (crotyl chloride) [591-97-9] YTKRILODNOEEPX-UHFFFAOYSA-N | $9.5\times10^{-4}$ | | Ebert et al. (2023) | ? | 318 |
| ($Z$)-1-chloro-2-butene $C_4H_7Cl$ (cis-1-chloro-2-butene) [4628-21-1] YTKRILODNOEEPX-IHWYPQMZSA-N | $1.2\times10^{-3}$ | 2800 3800 2800 | Bakierowska and Trzeszczyński (2003) Kühne et al. (2005) Kühne et al. (2005) | M Q ? | |
| ($E$)-1-chloro-2-butene $C_4H_7Cl$ (trans-1-chloro-2-butene) [4894-61-5] YTKRILODNOEEPX-NSCUHMNNSA-N | $3.1\times10^{-3}$ | 3000 3800 3000 | Bakierowska and Trzeszczyński (2003) Kühne et al. (2005) Kühne et al. (2005) | M Q ? | |
| 1,3-dichloro-2-butene $C_4H_6Cl_2$ [926-57-8] WLIADPFXSACYLS-UHFFFAOYSA-N | $2.6\times10^{-4}$ | | HSDB (2015) | Q | 99 |
| 1,4-dichloro-2-butene $C_4H_6Cl_2$ [764-41-0] FQDIANVAWVHZIR-UHFFFAOYSA-N | $1.2\times10^{-3}$ $1.7\times10^{-2}$ $4.2\times10^{-3}$ $8.9\times10^{-3}$ | | Duchowicz et al. (2020) HSDB (2015) Duchowicz et al. (2020) Modarresi et al. (2007) | V V Q Q | 186 67 |
| ($Z$)-1,4-dichloro-2-butene $C_4H_6Cl_2$ [1476-11-5] FQDIANVAWVHZIR-UPHRSURJSA-N | $3.0\times10^{-2}$ $1.0\times10^{-2}$ $8.5\times10^{-3}$ $8.2\times10^{-3}$ $4.2\times10^{-3}$ | 9400 | Hiatt (2013) Albanese et al. (1987) Duchowicz et al. (2020) HSDB (2015) Duchowicz et al. (2020) | M M V V Q | 186 |
| ($E$)-1,4-dichloro-2-butene $C_4H_6Cl_2$ [110-57-6] FQDIANVAWVHZIR-OWOJBTEDSA-N | $3.5\times10^{-2}$ $1.7\times10^{-2}$ $1.5\times10^{-2}$ $1.5\times10^{-2}$ $4.2\times10^{-3}$ $7.2\times10^{-2}$ $8.9\times10^{-3}$ | 6600 | Hiatt (2013) Albanese et al. (1987) Duchowicz et al. (2020) HSDB (2015) Duchowicz et al. (2020) Hilal et al. (2008) Modarresi et al. (2007) | M M V V Q Q Q | 186 67 |
| 3,4-dichloro-1-butene $C_4H_6Cl_2$ [760-23-6] XVEASTGLHPVZNA-UHFFFAOYSA-N | $1.2\times10^{-3}$ $1.1\times10^{-3}$ $6.4\times10^{-3}$ | | Duchowicz et al. (2020) HSDB (2015) Duchowicz et al. (2020) | V V Q | 186 |



Table A6.1: Chlorocarbons (C, H, Cl) (... continued)

| Substance<br>Formula<br>(Trivial Name)<br>[CAS Registry Number]<br>InChIKey | $H_s^{cp}$<br>(at $T^\ominus$)<br>$\left[\dfrac{\mathrm{mol}}{\mathrm{m^3\,Pa}}\right]$ | $\dfrac{\mathrm{d\ln} H_s^{cp}}{\mathrm{d}(1/T)}$<br><br>[K] | Reference | Type | Note |
|---|---|---|---|---|---|
| 1-chloro-2-methylpropene<br>$C_4H_7Cl$<br>(dimethylvinyl chloride)<br>[513-37-1]<br>KWISWUFGPUHDRY-UHFFFAOYSA-N | $5.2\times10^{-4}$ | | HSDB (2015)<br>Haynes (2014) | V<br>W | <br>705 |
| 2-chloro-1,3-butadiene<br>$C_4H_5Cl$<br>[126-99-8]<br>YACLQRRMGMJLJV-UHFFFAOYSA-N | $4.7\times10^{-2}$<br>$1.8\times10^{-4}$ | | Mackay et al. (1993)<br>HSDB (2015) | V<br>Q | <br>99 |
| hexachlorobutadiene<br>$CCl_2CClCClCCl_2$<br>[87-68-3]<br>RWNKSTSCBHKHTB-UHFFFAOYSA-N | $5.6\times10^{-4}$<br>$8.3\times10^{-4}$<br>$2.3\times10^{-3}$<br>$6.2\times10^{-4}$<br>$7.0\times10^{-4}$<br>$2.3\times10^{-3}$<br>$9.6\times10^{-4}$<br>$4.0\times10^{-4}$<br>$6.1\times10^{-4}$<br>$6.5\times10^{-4}$<br>$9.1\times10^{-4}$<br>$3.8\times10^{-4}$<br>$4.2\times10^{-4}$<br>$9.8\times10^{-4}$<br>$9.7\times10^{-4}$<br>$9.6\times10^{-4}$<br>$9.4\times10^{-4}$<br>$9.6\times10^{-4}$<br>$9.0\times10^{-4}$<br>$5.0\times10^{-4}$<br>$2.3\times10^{-3}$<br>$1.7\times10^{-2}$<br>$4.3\times10^{-4}$<br>$6.2\times10^{-4}$<br>$7.6\times10^{-3}$<br><br>$2.7\times10^{-4}$<br><br>$4.2\times10^{-4}$ | 5500<br>3100<br>6200<br>4900<br>2500<br><br><br><br><br><br><br><br><br>4600<br><br><br><br><br><br><br><br><br><br><br><br>5300<br><br>3500<br> | Brockbank (2013)<br>Fogg and Sangster (2003)<br>Hiatt (2013)<br>Dewulf et al. (1999)<br>Kondoh and Nakajima (1997)<br>Oliver (1985)<br>Warner et al. (1980)<br>Pearson and McConnell (1975)<br>Mackay et al. (2006b)<br>Mackay et al. (1993)<br>Ballschmiter and Wittlinger (1991)<br>Warner et al. (1980)<br>Yaws (2003)<br>Goldstein (1982)<br>Hilal et al. (2008)<br>Horvath and Getzen (1999)<br>Ryan et al. (1988)<br>Shen (1982)<br>Zhang et al. (2010)<br>Zhang et al. (2010)<br>Zhang et al. (2010)<br>Zhang et al. (2010)<br>Gharagheizi et al. (2010)<br>Hilal et al. (2008)<br>Modarresi et al. (2007)<br>Kühne et al. (2005)<br>Yao et al. (2002)<br>Kühne et al. (2005)<br>Yaws (1999) | L<br>L<br>M<br>M<br>M<br>M<br>M<br>M<br>V<br>V<br>V<br>V<br>X<br>X<br>C<br>C<br>C<br>C<br>Q<br>Q<br>Q<br>Q<br>Q<br>Q<br>Q<br>Q<br>Q<br>?<br>? | 1<br><br><br><br><br><br><br>649, 12<br><br><br><br><br>237<br>298<br><br><br><br><br>287, 288<br>287, 289<br>287, 290<br>287, 291<br>246<br><br>67<br><br>229<br><br>21 |
| hexachlorocyclopentadiene<br>$C_5Cl_6$<br>[77-47-4]<br>VUNCWTMEJYMOOR-UHFFFAOYSA-N | $3.7\times10^{-4}$<br>$6.0\times10^{-4}$<br>$6.1\times10^{-4}$<br>$6.0\times10^{-4}$<br>$6.2\times10^{-4}$<br>$2.7\times10^{-4}$<br>$6.0\times10^{-4}$<br>$2.7\times10^{-4}$ | <br><br><br><br><br><br>1500<br> | Wolfe et al. (1982)<br>Warner et al. (1980)<br>Mackay et al. (2006b)<br>Mackay et al. (1993)<br>Wolfe et al. (1982)<br>Warner et al. (1980)<br>Goldstein (1982)<br>Ryan et al. (1988) | M<br>M<br>V<br>V<br>V<br>V<br>X<br>C | <br><br><br><br><br><br>298<br> |



Table A6.1: Chlorocarbons (C, H, Cl) (...continued)

| Substance Formula (Trivial Name) [CAS Registry Number] InChIKey | $H_s^{cp}$ (at $T^{\ominus}$) $\left[\dfrac{\text{mol}}{\text{m}^3\,\text{Pa}}\right]$ | $\dfrac{\text{d}\ln H_s^{cp}}{\text{d}(1/T)}$ [K] | Reference | Type | Note |
|---|---|---|---|---|---|
| | $6.0\times10^{-4}$ | | Shen (1982) | C | |
| | $4.6\times10^{-3}$ | | Zhang et al. (2010) | Q | 287, 288 |
| | $5.3\times10^{-3}$ | | Zhang et al. (2010) | Q | 287, 289 |
| | $1.4\times10^{-2}$ | | Zhang et al. (2010) | Q | 287, 290 |
| | $1.6\times10^{-2}$ | | Zhang et al. (2010) | Q | 287, 291 |
| | $2.3\times10^{-3}$ | | Hilal et al. (2008) | Q | |
| | $2.2\times10^{-2}$ | | Modarresi et al. (2007) | Q | 67 |
| | $3.7\times10^{-4}$ | | Yaffe et al. (2003) | Q | 248, 249 |
| | $2.4\times10^{-3}$ | | Katritzky et al. (1998) | Q | |
| chlordane $C_{10}H_6Cl_8$ [57-74-9] BIWJNBZANLAXMG-UHFFFAOYSA-N | $1.8\times10^{-1}$ | | Fendinger et al. (1989) | M | 72 |
| | $1.2\times10^{-1}$ | | Fendinger et al. (1989) | M | 645 |
| | $2.1\times10^{-1}$ | | Warner et al. (1980) | M | |
| | $1.1\times10^{-1}$ | | Suntio et al. (1988) | V | 12 |
| | $1.1\times10^{-3}$ | | Barcelo and Hennion (1997) | X | 567 |
| | $2.0\times10^{-1}$ | | Suntio et al. (1988) | C | |
| | $1.1\times10^{-1}$ | | Suntio et al. (1988) | C | |
| | $1.0\times10^{-1}$ | | Ryan et al. (1988) | C | |
| | $2.1\times10^{-1}$ | | Shen (1982) | C | |
| | $1.3\times10^{-1}$ | | Keshavarz et al. (2022) | Q | |
| | $3.6\times10^{-2}$ | | Duchowicz et al. (2020) | Q | 184 |
| | $1.4\times10^{-1}$ | | Zhang et al. (2010) | Q | 287, 288 |
| | $4.8\times10^{-2}$ | | Zhang et al. (2010) | Q | 287, 289 |
| | $2.4\times10^{1}$ | | Zhang et al. (2010) | Q | 287, 290 |
| | $1.5$ | | Zhang et al. (2010) | Q | 287, 291 |
| | $2.1\times10^{-4}$ | | Goodarzi et al. (2010) | Q | 568, 569 |
| | $5.3\times10^{-2}$ | | Hilal et al. (2008) | Q | |
| | $2.0\times10^{-1}$ | | Duchowicz et al. (2020) | ? | 185, 21 |
| *cis*-chlordane $C_{10}H_6Cl_8$ ($\alpha$-chlordane) [5103-71-9] BIWJNBZANLAXMG-KMMBHOGFSA-N | $1.7\times10^{-1}$ | | Shen and Wania (2005) | L | 366 |
| | $1.8\times10^{-1}$ | | Shen and Wania (2005) | L | 367 |
| | $3.7\times10^{-2}$ | 4100 | Jantunen and Bidleman (2006) | M | |
| | $1.5\times10^{-1}$ | 6100 | Cetin et al. (2006) | M | |
| | $1.1\times10^{-2}$ | | Atlas et al. (1982) | M | 679 |
| | $2.8\times10^{-2}$ | | Duchowicz et al. (2020) | V | 186 |
| | | | Mackay et al. (2006d) | V | 558 |
| | $4.8\times10^{-3}$ | 7300 | Paasivirta et al. (1999) | T | |
| | $3.6\times10^{-2}$ | | Duchowicz et al. (2020) | Q | |
| *trans*-chlordane $C_{10}H_6Cl_8$ ($\beta$-chlordane) [5103-74-2] BIWJNBZANLAXMG-OESJLNMISA-N | $1.7\times10^{-1}$ | | Shen and Wania (2005) | L | 366 |
| | $1.5\times10^{-1}$ | | Shen and Wania (2005) | L | 367 |
| | $3.4\times10^{-2}$ | 3500 | Jantunen and Bidleman (2006) | M | |
| | $6.3\times10^{-2}$ | 7600 | Cetin et al. (2006) | M | |
| | $7.4\times10^{-3}$ | | Atlas et al. (1982) | M | 679 |
| | $2.0\times10^{-2}$ | | Duchowicz et al. (2020) | V | 186 |
| | | | Mackay et al. (2006d) | V | 558 |
| | $3.6\times10^{-3}$ | 7100 | Paasivirta et al. (1999) | T | |
| | $3.6\times10^{-2}$ | | Duchowicz et al. (2020) | Q | |
| | $1.0$ | | Modarresi et al. (2007) | Q | 67 |





Table A6.1: Chlorocarbons (C, H, Cl) (. . . continued)

| Substance Formula (Trivial Name) [CAS Registry Number] InChIKey | $H_s^{cp}$ (at $T^{\ominus}$) $\left[\dfrac{\text{mol}}{\text{m}^3\,\text{Pa}}\right]$ | $\dfrac{\text{d}\ln H_s^{cp}}{\text{d}(1/T)}$ [K] | Reference | Type | Note |
|---|---|---|---|---|---|
| $\gamma$-chlordane $C_{10}H_6Cl_8$ [5566-34-7] JBZJEPYXXVKOKF-UHFFFAOYSA-N | $6.3\times10^{-2}$ | | Ebert et al. (2023) | ? | 579 |
| *cis*-nonachlor $C_{10}H_5Cl_9$ [5103-73-1] OCHOKXCPKDPNQU-BBXWSCHTSA-N | 1.4 | 5100 | Cetin et al. (2006) | M | |
| *trans*-nonachlor $C_{10}H_5Cl_9$ [39765-80-5] OCHOKXCPKDPNQU-FLVMBEMLSA-N | $3.1\times10^{-2}$ $8.8\times10^{-2}$ $7.9\times10^{-4}$ | 4800 8000 7600 | Jantunen and Bidleman (2006) Cetin et al. (2006) Paasivirta et al. (1999) | M M T | |
| chlorobenzene $C_6H_5Cl$ [108-90-7] MVPPADPHJFYWMZ-UHFFFAOYSA-N | $2.4\times10^{-3}$ $2.6\times10^{-3}$ $2.7\times10^{-3}$ $2.7\times10^{-3}$ $2.9\times10^{-3}$ $2.0\times10^{-3}$ $3.7\times10^{-3}$ $1.7\times10^{-3}$ $2.4\times10^{-3}$ $1.5\times10^{-3}$ $2.5\times10^{-3}$ $1.9\times10^{-3}$ $3.6\times10^{-3}$ $3.4\times10^{-3}$ $2.7\times10^{-3}$ $3.2\times10^{-3}$ $3.5\times10^{-3}$ $3.0\times10^{-3}$ $1.9\times10^{-3}$ $2.9\times10^{-3}$ $3.0\times10^{-3}$ $2.6\times10^{-3}$ $3.1\times10^{-3}$ $2.5\times10^{-3}$ $3.0\times10^{-3}$ $2.4\times10^{-3}$ $2.5\times10^{-3}$ $2.9\times10^{-3}$ $3.1\times10^{-3}$ $3.2\times10^{-3}$ $3.0\times10^{-3}$ $2.9\times10^{-3}$ $2.5\times10^{-3}$ | 3700 3600 3800 3800 3800 4400 1300 2300 4300 2900 2900 3100 4700 2700 3500 4200 | Schwardt et al. (2021) Brockbank (2013) Staudinger and Roberts (2001) Staudinger and Roberts (1996) Mackay and Shiu (1981) Schwardt et al. (2021) Hiatt (2013) Lau et al. (2010) Li et al. (2008) Lei et al. (2004) Dewulf et al. (1999) Ryu and Park (1999) Dohnal and Hovorka (1999) de Wolf and Lieder (1998) Welke et al. (1998) Shiu and Mackay (1997) Hovorka and Dohnal (1997) Kondoh and Nakajima (1997) Park et al. (1997) Ramachandran et al. (1996) Khalfaoui and Newsham (1994b) Hoff et al. (1993) Ettre et al. (1993) Li and Carr (1993) Cooling et al. (1992) Bissonette et al. (1990) Ashworth et al. (1988) Hellmann (1987) Yurteri et al. (1987) Mackay and Shiu (1981) Leighton and Calo (1981) Ervin et al. (1980) Warner et al. (1980) | L L L L L M M M M M M M M M M M M M M M M M M M M M M M M M M M | 1, 706 1 707 33, 11 327 12 87 12 708 11 709 278 87 12 |



Table A6.1: Chlorocarbons (C, H, Cl) (. . . continued)

| Substance<br>Formula<br>(Trivial Name)<br>[CAS Registry Number]<br>InChIKey | $H_s^{cp}$<br>(at $T^{\ominus}$)<br>$\left[\dfrac{\text{mol}}{\text{m}^3\,\text{Pa}}\right]$ | $\dfrac{\text{d}\ln H_s^{cp}}{\text{d}(1/T)}$<br><br>[K] | Reference | Type | Note |
|---|---|---|---|---|---|
| | $2.6\times10^{-3}$ | | Mackay et al. (1979) | M | |
| | $1.6\times10^{-3}$ | | Sato and Nakajima (1979b) | M | 14 |
| | $2.8\times10^{-3}$ | 4900 | Hartkopf and Karger (1973) | M | |
| | $2.7\times10^{-3}$ | | Mackay et al. (2006b) | V | |
| | $2.9\times10^{-3}$ | 2400 | Fogg and Sangster (2003) | V | |
| | $2.7\times10^{-3}$ | | Shiu and Mackay (1997) | V | |
| | $2.8\times10^{-3}$ | | Park et al. (1997) | V | |
| | $2.9\times10^{-3}$ | | Lide and Frederikse (1995) | V | |
| | $2.7\times10^{-3}$ | | Mackay et al. (1993) | V | |
| | $2.7\times10^{-3}$ | | Mackay et al. (1992a) | V | |
| | $2.5\times10^{-3}$ | | Hwang et al. (1992) | V | |
| | $2.7\times10^{-3}$ | | Bobra et al. (1985) | V | |
| | $2.7\times10^{-3}$ | | Yoshida et al. (1983) | V | |
| | $2.7\times10^{-3}$ | | Cabani et al. (1981) | V | |
| | $2.7\times10^{-3}$ | | Warner et al. (1980) | V | |
| | $2.2\times10^{-3}$ | | Hine and Mookerjee (1975) | V | |
| | $2.7\times10^{-3}$ | | Mackay et al. (1979) | T | |
| | $2.2\times10^{-3}$ | | Yaws (2003) | X | 237 |
| | $2.5\times10^{-3}$ | 2100 | Goldstein (1982) | X | 298 |
| | $2.7\times10^{-3}$ | | Schüürmann (2000) | C | 21 |
| | $2.7\times10^{-3}$ | | Ryan et al. (1988) | C | |
| | $2.5\times10^{-3}$ | | Shen (1982) | C | |
| | $2.4\times10^{-2}$ | | Hayer et al. (2022) | Q | 20 |
| | $7.4\times10^{-3}$ | | Keshavarz et al. (2022) | Q | |
| | $8.6\times10^{-3}$ | | Duchowicz et al. (2020) | Q | |
| | $1.5\times10^{-2}$ | | Gharagheizi et al. (2012) | Q | |
| | $4.9\times10^{-3}$ | | Raventos-Duran et al. (2010) | Q | 271, 243 |
| | $3.1\times10^{-3}$ | | Raventos-Duran et al. (2010) | Q | 244 |
| | $2.5\times10^{-3}$ | | Raventos-Duran et al. (2010) | Q | 245 |
| | $1.5\times10^{-3}$ | | Gharagheizi et al. (2010) | Q | 246 |
| | $4.0\times10^{-3}$ | | Hilal et al. (2008) | Q | |
| | $8.6\times10^{-3}$ | | Modarresi et al. (2007) | Q | 67 |
| | | 4000 | Kühne et al. (2005) | Q | |
| | $2.9\times10^{-3}$ | | Yaffe et al. (2003) | Q | 248, 249 |
| | $2.8\times10^{-3}$ | | Delgado and Alderete (2002) | Q | |
| | $3.6\times10^{-3}$ | | Yao et al. (2002) | Q | 229 |
| | $4.0\times10^{-3}$ | | English and Carroll (2001) | Q | 230, 231 |
| | $1.7\times10^{-3}$ | | Katritzky et al. (1998) | Q | |
| | $1.5\times10^{-3}$ | | Myrdal and Yalkowsky (1994) | Q | |
| | $2.2\times10^{-3}$ | | Suzuki et al. (1992) | Q | 232 |
| | $4.2\times10^{-3}$ | | Nirmalakhandan and Speece (1988) | Q | |
| | $1.8\times10^{-3}$ | | Arbuckle (1983) | Q | |
| | $3.2\times10^{-3}$ | | Duchowicz et al. (2020) | ? | 185, 21 |
| | | 4000 | Kühne et al. (2005) | ? | |
| | $2.2\times10^{-3}$ | | Yaws (1999) | ? | 21 |
| | $1.6\times10^{-3}$ | | Abraham and Weathersby (1994) | ? | 21 |
| | $2.6\times10^{-3}$ | | Mackay et al. (1993) | ? | |



Table A6.1: Chlorocarbons (C, H, Cl) (... continued)

| Substance<br>Formula<br>(Trivial Name)<br>[CAS Registry Number]<br>InChIKey | $H_s^{cp}$<br>(at $T^{\ominus}$)<br>$\left[\dfrac{\text{mol}}{\text{m}^3\,\text{Pa}}\right]$ | $\dfrac{\text{d}\ln H_s^{cp}}{\text{d}(1/T)}$<br><br>[K] | Reference | Type | Note |
|---|---|---|---|---|---|
| | $2.2\times10^{-3}$ | | Yaws and Yang (1992) | ? | 21 |
| | $2.8\times10^{-3}$ | | Abraham et al. (1990) | ? | |
| | $3.8\times10^{-3}$ | | Mackay and Yeun (1983) | ? | |
| chlorobenzene-d5<br>$C_6D_5Cl$<br>[3114-55-4]<br>MVPPADPHJFYWMZ-RALIUCGRSA-N | $3.6\times10^{-3}$ | 4500 | Hiatt (2013) | M | |
| 1,2-dichlorobenzene<br>$C_6H_4Cl_2$<br>($o$-dichlorobenzene)<br>[95-50-1]<br>RFFLAFLAYFXFSW-UHFFFAOYSA-N | $5.6\times10^{-3}$ | 3700 | Schwardt et al. (2021) | L | 1 |
| | $5.9\times10^{-3}$ | 5200 | Brockbank (2013) | L | 1, 710 |
| | $6.8\times10^{-3}$ | 5300 | Fogg and Sangster (2003) | L | 711 |
| | $5.4\times10^{-3}$ | 5900 | Staudinger and Roberts (2001) | L | |
| | $5.4\times10^{-3}$ | 5900 | Staudinger and Roberts (1996) | L | |
| | $5.3\times10^{-3}$ | | Mackay and Shiu (1981) | L | |
| | $8.0\times10^{-3}$ | 4200 | Hiatt (2013) | M | |
| | $6.3\times10^{-3}$ | | Li et al. (2008) | M | |
| | $4.7\times10^{-3}$ | | Ryu and Park (1999) | M | |
| | $5.1\times10^{-3}$ | | Shiu and Mackay (1997) | M | |
| | $7.2\times10^{-3}$ | | Hovorka and Dohnal (1997) | M | 12 |
| | $6.2\times10^{-3}$ | 5000 | Kondoh and Nakajima (1997) | M | |
| | $4.9\times10^{-3}$ | 4400 | Park et al. (1997) | M | |
| | $4.8\times10^{-3}$ | | Li and Carr (1993) | M | |
| | $3.5\times10^{-3}$ | | Yu (1992) | M | 12 |
| | $4.9\times10^{-3}$ | 5100 | Bissonette et al. (1990) | M | |
| | $5.3\times10^{-3}$ | 1400 | Ashworth et al. (1988) | M | 42, 278 |
| | $8.2\times10^{-3}$ | | Oliver (1985) | M | |
| | $5.9\times10^{-3}$ | 6700 | Gossett et al. (1985) | M | |
| | $5.2\times10^{-3}$ | | Mackay and Shiu (1981) | M | |
| | $5.1\times10^{-3}$ | | Warner et al. (1980) | M | |
| | $3.5\times10^{-3}$ | | Sato and Nakajima (1979b) | M | 14 |
| | $5.6\times10^{-3}$ | | Mackay et al. (2006b) | V | |
| | $4.1\times10^{-3}$ | | Shiu and Mackay (1997) | V | |
| | $8.6\times10^{-3}$ | | Park et al. (1997) | V | |
| | $8.3\times10^{-3}$ | | Lide and Frederikse (1995) | V | |
| | $4.1\times10^{-3}$ | | Mackay et al. (1992a) | V | |
| | $6.0\times10^{-3}$ | | Hwang et al. (1992) | V | |
| | $4.1\times10^{-3}$ | | Bobra et al. (1985) | V | |
| | $4.9\times10^{-3}$ | | Warner et al. (1980) | V | |
| | $4.0\times10^{-3}$ | | Hine and Mookerjee (1975) | V | |
| | $3.5\times10^{-3}$ | | Yaws (2003) | X | 237 |
| | $5.2\times10^{-3}$ | 2800 | Goldstein (1982) | X | 298 |
| | $5.2\times10^{-3}$ | | Schüürmann (2000) | C | 21 |
| | $2.7\times10^{-3}$ | | Ryan et al. (1988) | C | |
| | $5.1\times10^{-3}$ | | Shen (1982) | C | |
| | $7.4\times10^{-3}$ | | Keshavarz et al. (2022) | Q | |
| | $1.5\times10^{-2}$ | | Duchowicz et al. (2020) | Q | 184 |
| | $4.0\times10^{-3}$ | | Li et al. (2014) | Q | 241 |



Table A6.1: Chlorocarbons (C, H, Cl) (. . . continued)

| Substance Formula (Trivial Name) [CAS Registry Number] InChIKey | $H_s^{cp}$ (at $T^{\ominus}$) $\left[\dfrac{\mathrm{mol}}{\mathrm{m}^3\,\mathrm{Pa}}\right]$ | $\dfrac{\mathrm{d}\ln H_s^{cp}}{\mathrm{d}(1/T)}$ [K] | Reference | Type | Note |
|---|---|---|---|---|---|
| | $4.7\times10^{-2}$ | | Gharagheizi et al. (2012) | Q | |
| | $6.2\times10^{-3}$ | | Raventos-Duran et al. (2010) | Q | 242, 243 |
| | $6.2\times10^{-3}$ | | Raventos-Duran et al. (2010) | Q | 244 |
| | $3.1\times10^{-3}$ | | Raventos-Duran et al. (2010) | Q | 245 |
| | $3.0\times10^{-3}$ | | Gharagheizi et al. (2010) | Q | 246 |
| | $8.2\times10^{-3}$ | | Hilal et al. (2008) | Q | |
| | $4.5\times10^{-3}$ | | Modarresi et al. (2007) | Q | 67 |
| | | 4400 | Kühne et al. (2005) | Q | |
| | $5.6\times10^{-3}$ | | Yaffe et al. (2003) | Q | 248, 249 |
| | $7.1\times10^{-3}$ | | Delgado and Alderete (2002) | Q | |
| | $8.0\times10^{-3}$ | | Yao et al. (2002) | Q | 229 |
| | $4.7\times10^{-3}$ | | English and Carroll (2001) | Q | 230, 231 |
| | $3.3\times10^{-3}$ | | Katritzky et al. (1998) | Q | |
| | $2.3\times10^{-3}$ | | Myrdal and Yalkowsky (1994) | Q | |
| | $8.4\times10^{-3}$ | | Nirmalakhandan and Speece (1988) | Q | |
| | $5.1\times10^{-3}$ | | Duchowicz et al. (2020) | ? | 185, 21 |
| | | 4800 | Kühne et al. (2005) | ? | |
| | $3.5\times10^{-3}$ | | Yaws (1999) | ? | 21 |
| | $3.6\times10^{-3}$ | | Abraham and Weathersby (1994) | ? | 21 |
| | $3.3\times10^{-3}$ | | Yaws and Yang (1992) | ? | 21 |
| | $5.1\times10^{-3}$ | | Abraham et al. (1990) | ? | |
| | $6.2\times10^{-3}$ | | Chiou et al. (1980) | ? | 79 |
| 1,2-dichlorobenzene-d4 $C_6D_4Cl_2$ ($o$-dichlorobenzene-d4) [2199-69-1] RFFLAFLAYFXFSW-RHQRLBAQSA-N | $8.2\times10^{-3}$ | 4200 | Hiatt (2013) | M | |
| 1,3-dichlorobenzene $C_6H_4Cl_2$ ($m$-dichlorobenzene) [541-73-1] ZPQOPVIELGIULI-UHFFFAOYSA-N | $3.1\times10^{-3}$ | 3700 | Schwardt et al. (2021) | L | 1 |
| | $3.1\times10^{-3}$ | 4400 | Brockbank (2013) | L | 1 |
| | $3.4\times10^{-3}$ | 4300 | Fogg and Sangster (2003) | L | |
| | $2.8\times10^{-3}$ | | Mackay and Shiu (1981) | L | |
| | $5.2\times10^{-3}$ | 4800 | Hiatt (2013) | M | |
| | $2.9\times10^{-3}$ | | Li et al. (2008) | M | |
| | $3.7\times10^{-3}$ | | de Wolf and Lieder (1998) | M | 87 |
| | $4.7\times10^{-3}$ | | Hovorka and Dohnal (1997) | M | 12 |
| | $3.8\times10^{-3}$ | 4400 | Kondoh and Nakajima (1997) | M | |
| | $3.4\times10^{-3}$ | | Hoff et al. (1993) | M | |
| | $3.0\times10^{-3}$ | 2600 | Ashworth et al. (1988) | M | 33, 278 |
| | $5.5\times10^{-3}$ | | Oliver (1985) | M | |
| | $3.8\times10^{-3}$ | | Warner et al. (1980) | M | |
| | $2.1\times10^{-3}$ | | Sato and Nakajima (1979b) | M | 14 |
| | $3.1\times10^{-3}$ | | Mackay et al. (2006b) | V | |
| | $2.7\times10^{-3}$ | | Shiu and Mackay (1997) | V | |
| | $5.6\times10^{-3}$ | | Lide and Frederikse (1995) | V | |
| | $2.7\times10^{-3}$ | | Mackay et al. (1992a) | V | |
| | $2.7\times10^{-3}$ | | Bobra et al. (1985) | V | |



Table A6.1: Chlorocarbons (C, H, Cl) (...continued)

| Substance<br>Formula<br>(Trivial Name)<br>[CAS Registry Number]<br>InChIKey | $H_s^{cp}$<br>(at $T^\ominus$)<br>$\left[\dfrac{\text{mol}}{\text{m}^3\,\text{Pa}}\right]$ | $\dfrac{\text{d}\ln H_s^{cp}}{\text{d}(1/T)}$<br><br>[K] | Reference | Type | Note |
|---|---|---|---|---|---|
| | $3.3\times10^{-3}$ | | Warner et al. (1980) | V | |
| | $2.1\times10^{-3}$ | | Hine and Mookerjee (1975) | V | |
| | $2.9\times10^{-3}$ | | Yaws (2003) | X | 237 |
| | $3.9\times10^{-3}$ | 2400 | Goldstein (1982) | X | 298 |
| | $3.7\times10^{-3}$ | | Ryan et al. (1988) | C | |
| | $3.8\times10^{-3}$ | | Shen (1982) | C | |
| | $7.4\times10^{-3}$ | | Keshavarz et al. (2022) | Q | |
| | $9.2\times10^{-3}$ | | Duchowicz et al. (2020) | Q | 299 |
| | $3.9\times10^{-2}$ | | Gharagheizi et al. (2012) | Q | |
| | $6.2\times10^{-3}$ | | Raventos-Duran et al. (2010) | Q | 271, 243 |
| | $4.9\times10^{-3}$ | | Raventos-Duran et al. (2010) | Q | 244 |
| | $3.1\times10^{-3}$ | | Raventos-Duran et al. (2010) | Q | 245 |
| | $3.0\times10^{-3}$ | | Gharagheizi et al. (2010) | Q | 246 |
| | $4.7\times10^{-3}$ | | Hilal et al. (2008) | Q | |
| | $3.2\times10^{-3}$ | | Modarresi et al. (2007) | Q | 67 |
| | | 4100 | Kühne et al. (2005) | Q | |
| | $3.8\times10^{-3}$ | | Yaffe et al. (2003) | Q | 248, 249 |
| | $4.5\times10^{-3}$ | | Delgado and Alderete (2002) | Q | |
| | $5.0\times10^{-3}$ | | Yao et al. (2002) | Q | 229, 267 |
| | $3.7\times10^{-3}$ | | English and Carroll (2001) | Q | 230, 274 |
| | $4.2\times10^{-3}$ | | Katritzky et al. (1998) | Q | |
| | $2.3\times10^{-3}$ | | Myrdal and Yalkowsky (1994) | Q | |
| | $8.4\times10^{-3}$ | | Nirmalakhandan and Speece (1988) | Q | |
| | $3.8\times10^{-3}$ | | Duchowicz et al. (2020) | ? | 185, 21 |
| | | 4500 | Kühne et al. (2005) | ? | |
| | $2.9\times10^{-3}$ | | Yaws (1999) | ? | 21 |
| | $2.2\times10^{-3}$ | | Abraham and Weathersby (1994) | ? | 21 |
| | $3.0\times10^{-3}$ | | Yaws and Yang (1992) | ? | 21 |
| | $2.7\times10^{-3}$ | | Abraham et al. (1990) | ? | |
| 1,4-dichlorobenzene<br>$C_6H_4Cl_2$<br>(p-dichlorobenzene)<br>[106-46-7]<br>OCJBOOLMMGQPQU-UHFFFAOYSA-N | $3.2\times10^{-3}$ | 3800 | Schwardt et al. (2021) | L | 1 |
| | $3.8\times10^{-3}$ | 6000 | Brockbank (2013) | L | 1 |
| | $4.5\times10^{-3}$ | 4400 | Fogg and Sangster (2003) | L | |
| | $6.3\times10^{-3}$ | | Mackay and Shiu (1981) | L | |
| | $5.8\times10^{-3}$ | 4600 | Hiatt (2013) | M | |
| | $3.3\times10^{-3}$ | | Li et al. (2008) | M | |
| | $2.5\times10^{-3}$ | | Chiang et al. (1998) | M | 12 |
| | $4.1\times10^{-3}$ | | Shiu and Mackay (1997) | M | |
| | $5.4\times10^{-3}$ | | Hovorka and Dohnal (1997) | M | 12 |
| | $4.7\times10^{-3}$ | 4800 | Kondoh and Nakajima (1997) | M | |
| | $3.1\times10^{-3}$ | 2700 | Ashworth et al. (1988) | M | 278 |
| | $5.2\times10^{-3}$ | | Yurteri et al. (1987) | M | 12 |
| | $6.6\times10^{-3}$ | | Oliver (1985) | M | |
| | $4.2\times10^{-3}$ | | Mackay and Shiu (1981) | M | |
| | $3.6\times10^{-3}$ | | Warner et al. (1980) | M | |
| | $4.1\times10^{-3}$ | | Mackay et al. (2006b) | V | |
| | $6.3\times10^{-3}$ | | Shiu and Mackay (1997) | V | |
| | $6.7\times10^{-3}$ | | Lide and Frederikse (1995) | V | |





Table A6.1: Chlorocarbons (C, H, Cl) (. . . continued)

| Substance Formula (Trivial Name) [CAS Registry Number] InChIKey | $H_s^{cp}$ (at $T^\ominus$) $\left[\dfrac{\text{mol}}{\text{m}^3\,\text{Pa}}\right]$ | $\dfrac{\text{d}\ln H_s^{cp}}{\text{d}(1/T)}$ [K] | Reference | Type | Note |
|---|---|---|---|---|---|
| | $6.3\times10^{-3}$ | | Mackay et al. (1992a) | V | |
| | $3.8\times10^{-3}$ | | Suntio et al. (1988) | V | 12 |
| | $5.8\times10^{-3}$ | | Bobra et al. (1985) | V | |
| | $2.2\times10^{-3}$ | | Hine and Mookerjee (1975) | V | |
| | $2.3\times10^{-3}$ | | Yaws (2003) | X | 237 |
| | $3.7\times10^{-3}$ | 2700 | Goldstein (1982) | X | 298 |
| | $4.1\times10^{-3}$ | | Schüürmann (2000) | C | 21 |
| | $4.1\times10^{-3}$ | | Ryan et al. (1988) | C | |
| | $3.6\times10^{-3}$ | | Shen (1982) | C | |
| | $7.4\times10^{-3}$ | | Keshavarz et al. (2022) | Q | |
| | $8.4\times10^{-3}$ | | Duchowicz et al. (2020) | Q | 184 |
| | $3.9\times10^{-2}$ | | Gharagheizi et al. (2012) | Q | |
| | $6.2\times10^{-3}$ | | Raventos-Duran et al. (2010) | Q | 271, 243 |
| | $6.2\times10^{-3}$ | | Raventos-Duran et al. (2010) | Q | 244 |
| | $3.1\times10^{-3}$ | | Raventos-Duran et al. (2010) | Q | 245 |
| | $3.0\times10^{-3}$ | | Gharagheizi et al. (2010) | Q | 246 |
| | $6.5\times10^{-3}$ | | Hilal et al. (2008) | Q | |
| | $3.0\times10^{-3}$ | | Modarresi et al. (2007) | Q | 67 |
| | | 4100 | Kühne et al. (2005) | Q | |
| | $4.1\times10^{-3}$ | | Yaffe et al. (2003) | Q | 248, 249 |
| | $4.1\times10^{-3}$ | | Delgado and Alderete (2002) | Q | |
| | $4.3\times10^{-3}$ | | Katritzky et al. (1998) | Q | |
| | $2.3\times10^{-3}$ | | Myrdal and Yalkowsky (1994) | Q | |
| | $8.4\times10^{-3}$ | | Nirmalakhandan and Speece (1988) | Q | |
| | $2.1\times10^{-3}$ | | Arbuckle (1983) | Q | |
| | $4.1\times10^{-3}$ | | Duchowicz et al. (2020) | ? | 185, 21 |
| | | 3700 | Kühne et al. (2005) | ? | |
| | $2.3\times10^{-3}$ | | Yaws and Yang (1992) | ? | 21 |
| | $3.8\times10^{-3}$ | | Abraham et al. (1990) | ? | |
| 1,2,3-trichlorobenzene $C_6H_3Cl_3$ [87-61-6] RELMFMZEBKVZJC-UHFFFAOYSA-N | $6.2\times10^{-3}$ | 4600 | Brockbank (2013) | L | 1 |
| | $1.5\times10^{-2}$ | 4800 | Hiatt (2013) | M | |
| | $6.3\times10^{-3}$ | 4600 | Brockbank et al. (2013) | M | |
| | $8.0\times10^{-3}$ | | Lee et al. (2012) | M | |
| | $3.6\times10^{-3}$ | 4200 | Dewulf et al. (1999) | M | |
| | $7.9\times10^{-3}$ | | Shiu and Mackay (1997) | M | |
| | $1.5\times10^{-2}$ | 7300 | Kondoh and Nakajima (1997) | M | |
| | $1.4\times10^{-2}$ | | ten Hulscher et al. (1992) | M | 12 |
| | $1.1\times10^{-2}$ | | Oliver (1985) | M | |
| | $7.9\times10^{-3}$ | | Mackay and Shiu (1981) | M | |
| | $4.1\times10^{-3}$ | | Mackay et al. (2006b) | V | |
| | $5.8\times10^{-3}$ | | Fogg and Sangster (2003) | V | |
| | $2.1\times10^{-3}$ | | Fogg and Sangster (2003) | V | |
| | $4.1\times10^{-3}$ | | Shiu and Mackay (1997) | V | |
| | $3.3\times10^{-3}$ | | Abraham et al. (1994a) | V | |
| | $4.1\times10^{-3}$ | | Mackay et al. (1992a) | V | |
| | $4.2\times10^{-3}$ | | Bobra et al. (1985) | V | |
| | $4.3\times10^{-3}$ | | Mackay and Shiu (1981) | V | |



Table A6.1: Chlorocarbons (C, H, Cl) (...continued)

| Substance<br>Formula<br>(Trivial Name)<br>[CAS Registry Number]<br>InChIKey | $H_s^{cp}$<br>(at $T^{\ominus}$)<br>$\left[\dfrac{\text{mol}}{\text{m}^3\,\text{Pa}}\right]$ | $\dfrac{\text{d}\ln H_s^{cp}}{\text{d}(1/T)}$<br><br>[K] | Reference | Type | Note |
|---|---|---|---|---|---|
| | $7.4\times10^{-3}$ | | Keshavarz et al. (2022) | Q | |
| | $2.1\times10^{-2}$ | | Duchowicz et al. (2020) | Q | 299 |
| | $5.0\times10^{-3}$ | | Abraham et al. (2019) | Q | |
| | $8.0\times10^{-3}$ | | Li et al. (2014) | Q | 241 |
| | $7.8\times10^{-3}$ | | Raventos-Duran et al. (2010) | Q | 271, 243 |
| | $6.2\times10^{-3}$ | | Raventos-Duran et al. (2010) | Q | 244 |
| | $4.9\times10^{-3}$ | | Raventos-Duran et al. (2010) | Q | 245 |
| | $4.5\times10^{-3}$ | | Zhang et al. (2010) | Q | 287, 288 |
| | $6.9\times10^{-3}$ | | Zhang et al. (2010) | Q | 287, 289 |
| | $1.6\times10^{-2}$ | | Zhang et al. (2010) | Q | 287, 290 |
| | $5.2\times10^{-3}$ | | Zhang et al. (2010) | Q | 287, 291 |
| | $8.0\times10^{-3}$ | | Hilal et al. (2008) | Q | |
| | $4.6\times10^{-3}$ | | Modarresi et al. (2007) | Q | 67 |
| | | 4800 | Kühne et al. (2005) | Q | |
| | $1.1\times10^{-2}$ | | Delgado and Alderete (2002) | Q | |
| | $3.4\times10^{-3}$ | | English and Carroll (2001) | Q | 230, 231 |
| | $4.4\times10^{-3}$ | | Katritzky et al. (1998) | Q | |
| | $3.5\times10^{-3}$ | | Myrdal and Yalkowsky (1994) | Q | |
| | $1.8\times10^{-2}$ | | Nirmalakhandan and Speece (1988) | Q | |
| | $7.9\times10^{-3}$ | | Duchowicz et al. (2020) | ? | 185, 21 |
| | | 4200 | Kühne et al. (2005) | ? | |
| 1,2,3-trichlorobenzene-d3<br>$C_6D_3Cl_3$<br>[3907-98-0]<br>SMNOERSLNYGGOU-UHFFFAOYSA-N | $1.5\times10^{-2}$ | 4600 | Hiatt (2013) | M | |
| 1,2,4-trichlorobenzene<br>$C_6H_3Cl_3$<br>[120-82-1]<br>PBKONEOXTCPAFI-UHFFFAOYSA-N | $3.9\times10^{-3}$ | 3100 | Schwardt et al. (2021) | L | 1 |
| | $4.2\times10^{-3}$ | 6400 | Brockbank (2013) | L | 1 |
| | $1.1\times10^{-2}$ | 5100 | Hiatt (2013) | M | |
| | $5.8\times10^{-3}$ | | Lee et al. (2012) | M | |
| | $2.4\times10^{-3}$ | 3500 | Dewulf et al. (1999) | M | 712 |
| | $2.7\times10^{-3}$ | | Ryu and Park (1999) | M | |
| | $6.5\times10^{-3}$ | 5500 | Kondoh and Nakajima (1997) | M | |
| | $9.9\times10^{-3}$ | | ten Hulscher et al. (1992) | M | 12 |
| | $4.6\times10^{-3}$ | 3900 | Ashworth et al. (1988) | M | 33, 278 |
| | $8.2\times10^{-3}$ | | Oliver (1985) | M | |
| | $7.0\times10^{-3}$ | | Warner et al. (1980) | M | |
| | | | Mackay et al. (2006b) | V | 683 |
| | $7.1\times10^{-3}$ | | Fogg and Sangster (2003) | V | |
| | $8.6\times10^{-3}$ | | Fogg and Sangster (2003) | V | |
| | $3.6\times10^{-3}$ | | Shiu and Mackay (1997) | V | |
| | $7.1\times10^{-3}$ | | Lide and Frederikse (1995) | V | |
| | $3.6\times10^{-3}$ | | Mackay et al. (1992a) | V | |
| | $4.8\times10^{-3}$ | | McLachlan et al. (1990) | V | 373 |
| | $3.6\times10^{-3}$ | | Bobra et al. (1985) | V | |
| | $2.5\times10^{-3}$ | | Yoshida et al. (1983) | V | |
| | $2.6\times10^{-3}$ | | Mackay and Shiu (1981) | V | |



Table A6.1: Chlorocarbons (C, H, Cl) (... continued)

| Substance<br>Formula<br>(Trivial Name)<br>[CAS Registry Number]<br>InChIKey | $H_s^{cp}$<br>(at $T^\ominus$)<br>$\left[\dfrac{\text{mol}}{\text{m}^3\,\text{Pa}}\right]$ | $\dfrac{\mathrm{d}\ln H_s^{cp}}{\mathrm{d}(1/T)}$<br><br>[K] | Reference | Type | Note |
|---|---|---|---|---|---|
| | $4.3\times10^{-3}$ | | Warner et al. (1980) | V | |
| | $3.3\times10^{-3}$ | | Yaws (2003) | X | 237 |
| | $7.0\times10^{-3}$ | | Goldstein (1982) | X | 446 |
| | $7.0\times10^{-3}$ | 2500 | Goldstein (1982) | X | 298 |
| | $6.9\times10^{-3}$ | | Meylan and Howard (1991) | C | |
| | $4.2\times10^{-4}$ | | Ryan et al. (1988) | C | |
| | $7.0\times10^{-3}$ | | Shen (1982) | C | |
| | $7.4\times10^{-3}$ | | Keshavarz et al. (2022) | Q | |
| | $1.2\times10^{-2}$ | | Duchowicz et al. (2020) | Q | |
| | $2.5\times10^{-3}$ | | Abraham et al. (2019) | Q | |
| | $4.5\times10^{-3}$ | | Zhang et al. (2010) | Q | 287, 288 |
| | $7.7\times10^{-3}$ | | Zhang et al. (2010) | Q | 287, 289 |
| | $1.5\times10^{-2}$ | | Zhang et al. (2010) | Q | 287, 290 |
| | $4.6\times10^{-3}$ | | Zhang et al. (2010) | Q | 287, 291 |
| | $2.9\times10^{-3}$ | | Gharagheizi et al. (2010) | Q | 246 |
| | $9.9\times10^{-3}$ | | Hilal et al. (2008) | Q | |
| | $4.2\times10^{-3}$ | | Modarresi et al. (2007) | Q | 67 |
| | | 4500 | Kühne et al. (2005) | Q | |
| | $2.4\times10^{-3}$ | | Yaffe et al. (2003) | Q | 248, 249 |
| | $6.7\times10^{-3}$ | | Delgado and Alderete (2002) | Q | |
| | $7.4\times10^{-3}$ | | Yao et al. (2002) | Q | 229 |
| | $3.4\times10^{-3}$ | | English and Carroll (2001) | Q | 230, 260 |
| | $6.4\times10^{-3}$ | | Katritzky et al. (1998) | Q | |
| | $1.6\times10^{-2}$ | | Nirmalakhandan et al. (1997) | Q | |
| | $3.5\times10^{-3}$ | | Myrdal and Yalkowsky (1994) | Q | |
| | $4.5\times10^{-3}$ | | Meylan and Howard (1991) | Q | |
| | $6.9\times10^{-3}$ | | Duchowicz et al. (2020) | ? | 185, 21 |
| | | 3200 | Kühne et al. (2005) | ? | |
| | $3.3\times10^{-3}$ | | Yaws (1999) | ? | 21 |
| 1,2,4-trichlorobenzene-d3<br>$C_6D_3Cl_3$<br>[2199-72-6]<br>PBKONEOXTCPAFI-CBYSEHNBSA-N | $9.8\times10^{-3}$ | 4600 | Hiatt (2013) | M | |
| 1,3,5-trichlorobenzene<br>$C_6H_3Cl_3$<br>[108-70-3]<br>XKEFYDZQGKAQCN-UHFFFAOYSA-N | $1.8\times10^{-3}$ | 4100 | Dewulf et al. (1999) | M | 713 |
| | $5.2\times10^{-3}$ | | ten Hulscher et al. (1992) | M | 12 |
| | $3.5\times10^{-2}$ | | Hellmann (1987) | M | 87 |
| | $5.2\times10^{-3}$ | | Oliver (1985) | M | |
| | | | Mackay et al. (2006b) | V | 683 |
| | $1.4\times10^{-3}$ | | Fogg and Sangster (2003) | V | |
| | $8.5\times10^{-4}$ | | Fogg and Sangster (2003) | V | |
| | $9.1\times10^{-4}$ | | Shiu and Mackay (1997) | V | |
| | $1.0\times10^{-2}$ | | Lide and Frederikse (1995) | V | |
| | $1.5\times10^{-3}$ | | Abraham et al. (1994a) | V | |
| | $9.1\times10^{-4}$ | | Mackay et al. (1992a) | V | |
| | $9.1\times10^{-4}$ | | Bobra et al. (1985) | V | |
| | $6.2\times10^{-3}$ | | Mackay and Shiu (1981) | V | |



Table A6.1: Chlorocarbons (C, H, Cl) (. . . continued)

| Substance Formula (Trivial Name) [CAS Registry Number] InChIKey | $H_s^{cp}$ (at $T^\ominus$) $\left[\dfrac{\text{mol}}{\text{m}^3\,\text{Pa}}\right]$ | $\dfrac{\text{d}\ln H_s^{cp}}{\text{d}(1/T)}$ [K] | Reference | Type | Note |
|---|---|---|---|---|---|
| | $7.4\times10^{-3}$ | | Keshavarz et al. (2022) | Q | |
| | $7.5\times10^{-3}$ | | Duchowicz et al. (2020) | Q | 184 |
| | $7.8\times10^{-3}$ | | Raventos-Duran et al. (2010) | Q | 242, 243 |
| | $4.9\times10^{-3}$ | | Raventos-Duran et al. (2010) | Q | 244 |
| | $4.9\times10^{-3}$ | | Raventos-Duran et al. (2010) | Q | 245 |
| | $4.6\times10^{-3}$ | | Hilal et al. (2008) | Q | |
| | $4.2\times10^{-3}$ | | Modarresi et al. (2007) | Q | 67 |
| | | 4200 | Kühne et al. (2005) | Q | |
| | $4.5\times10^{-3}$ | | Yaffe et al. (2003) | Q | 248, 249 |
| | $4.6\times10^{-3}$ | | Delgado and Alderete (2002) | Q | |
| | $1.6\times10^{-2}$ | | Nirmalakhandan et al. (1997) | Q | |
| | $3.5\times10^{-3}$ | | Myrdal and Yalkowsky (1994) | Q | |
| | $4.5\times10^{-3}$ | | Meylan and Howard (1991) | Q | |
| | $1.8\times10^{-3}$ | | Rumble (2021) | ? | 714 |
| | $5.2\times10^{-3}$ | | Duchowicz et al. (2020) | ? | 185, 21 |
| | | 4400 | Kühne et al. (2005) | ? | |
| 1,2,3,4-tetrachlorobenzene $C_6H_2Cl_4$ [634-66-2] GBDZXPJXOMHESU-UHFFFAOYSA-N | $3.5\times10^{-3}$ | | Ryu and Park (1999) | M | |
| | $1.3\times10^{-2}$ | 4800 | ten Hulscher et al. (1992) | M | |
| | $5.7\times10^{-2}$ | | Hellmann (1987) | M | 87 |
| | $1.4\times10^{-2}$ | | Oliver (1985) | M | |
| | $9.0\times10^{-3}$ | | Mackay et al. (2006b) | V | |
| | $6.9\times10^{-3}$ | | Shiu and Mackay (1997) | V | |
| | $6.9\times10^{-3}$ | | Mackay et al. (1992a) | V | |
| | $5.8\times10^{-3}$ | | McLachlan et al. (1990) | V | 373 |
| | $6.9\times10^{-3}$ | | Bobra et al. (1985) | V | |
| | $3.8\times10^{-3}$ | | Mackay and Shiu (1981) | V | |
| | $7.4\times10^{-3}$ | | Keshavarz et al. (2022) | Q | |
| | $2.0\times10^{-2}$ | | Duchowicz et al. (2020) | Q | 299 |
| | $7.8\times10^{-3}$ | | Raventos-Duran et al. (2010) | Q | 242, 243 |
| | $7.8\times10^{-3}$ | | Raventos-Duran et al. (2010) | Q | 244 |
| | $6.2\times10^{-3}$ | | Raventos-Duran et al. (2010) | Q | 245 |
| | $6.1\times10^{-3}$ | | Zhang et al. (2010) | Q | 287, 288 |
| | $7.7\times10^{-3}$ | | Zhang et al. (2010) | Q | 287, 289 |
| | $2.1\times10^{-2}$ | | Zhang et al. (2010) | Q | 287, 290 |
| | $4.6\times10^{-3}$ | | Zhang et al. (2010) | Q | 287, 291 |
| | $8.6\times10^{-3}$ | | Hilal et al. (2008) | Q | |
| | $5.7\times10^{-3}$ | | Modarresi et al. (2007) | Q | 67 |
| | | 5200 | Kühne et al. (2005) | Q | |
| | $1.1\times10^{-2}$ | | Delgado and Alderete (2002) | Q | |
| | $4.2\times10^{-3}$ | | English and Carroll (2001) | Q | 230, 231 |
| | $5.7\times10^{-3}$ | | Myrdal and Yalkowsky (1994) | Q | |
| | $6.1\times10^{-3}$ | | Meylan and Howard (1991) | Q | |
| | $1.3\times10^{-2}$ | | Duchowicz et al. (2020) | ? | 185, 21 |
| | | 4500 | Kühne et al. (2005) | ? | |



Table A6.1: Chlorocarbons (C, H, Cl) (...continued)

| Substance Formula (Trivial Name) [CAS Registry Number] InChIKey | $H_s^{cp}$ (at $T^\ominus$) $\left[\dfrac{\mathrm{mol}}{\mathrm{m^3\,Pa}}\right]$ | $\dfrac{\mathrm{d}\ln H_s^{cp}}{\mathrm{d}(1/T)}$ [K] | Reference | Type | Note |
|---|---|---|---|---|---|
| 1,2,3,5-tetrachlorobenzene | $6.3\times10^{-3}$ | | Shiu and Mackay (1997) | M | |
| $C_6H_2Cl_4$ | $1.0\times10^{-2}$ | | ten Hulscher et al. (1992) | M | 12 |
| [634-90-2] | $6.3\times10^{-3}$ | | Mackay and Shiu (1981) | M | |
| QZYNWJQFTJXIRN-UHFFFAOYSA-N | $1.7\times10^{-3}$ | | Mackay et al. (2006b) | V | |
| | $2.1\times10^{-3}$ | | Fogg and Sangster (2003) | V | |
| | $1.8\times10^{-3}$ | | Fogg and Sangster (2003) | V | |
| | $1.7\times10^{-3}$ | | Shiu and Mackay (1997) | V | |
| | $1.7\times10^{-3}$ | | Mackay et al. (1992a) | V | |
| | $1.7\times10^{-3}$ | | Bobra et al. (1985) | V | |
| | $1.7\times10^{-3}$ | | Mackay and Shiu (1981) | V | |
| | $6.3\times10^{-3}$ | | Meylan and Howard (1991) | C | |
| | $7.4\times10^{-3}$ | | Keshavarz et al. (2022) | Q | |
| | $1.2\times10^{-2}$ | | Duchowicz et al. (2020) | Q | |
| | $1.7\times10^{-3}$ | | Li et al. (2014) | Q | 241 |
| | $7.8\times10^{-3}$ | | Raventos-Duran et al. (2010) | Q | 242, 243 |
| | $6.2\times10^{-3}$ | | Raventos-Duran et al. (2010) | Q | 244 |
| | $6.2\times10^{-3}$ | | Raventos-Duran et al. (2010) | Q | 245 |
| | $7.7\times10^{-3}$ | | Hilal et al. (2008) | Q | |
| | $5.4\times10^{-3}$ | | Modarresi et al. (2007) | Q | 67 |
| | $7.1\times10^{-3}$ | | Delgado and Alderete (2002) | Q | |
| | $4.2\times10^{-3}$ | | English and Carroll (2001) | Q | 230, 274 |
| | $3.2\times10^{-2}$ | | Nirmalakhandan et al. (1997) | Q | |
| | $5.7\times10^{-3}$ | | Myrdal and Yalkowsky (1994) | Q | |
| | $6.1\times10^{-3}$ | | Meylan and Howard (1991) | Q | |
| | $2.4\times10^{-2}$ | | Nirmalakhandan and Speece (1988) | Q | |
| | $6.2\times10^{-3}$ | | Duchowicz et al. (2020) | ? | 185, 21 |
| 1,2,4,5-tetrachlorobenzene | $1.8\times10^{-2}$ | | McPhedran et al. (2013) | M | |
| $C_6H_2Cl_4$ | $6.6\times10^{-3}$ | | Lee et al. (2012) | M | |
| [95-94-3] | $9.9\times10^{-3}$ | | Oliver (1985) | M | |
| JHBKHLUZVFWLAG-UHFFFAOYSA-N | $8.2\times10^{-3}$ | | Mackay et al. (2006b) | V | |
| | $2.8\times10^{-4}$ | | Fogg and Sangster (2003) | V | |
| | $1.1\times10^{-3}$ | | Fogg and Sangster (2003) | V | |
| | $8.2\times10^{-3}$ | | Shiu and Mackay (1997) | V | |
| | $8.2\times10^{-3}$ | | Mackay et al. (1992a) | V | |
| | $8.2\times10^{-3}$ | | Bobra et al. (1985) | V | |
| | $3.8\times10^{-3}$ | | Mackay and Shiu (1981) | V | |
| | $7.8\times10^{-3}$ | | Raventos-Duran et al. (2010) | Q | 242, 243 |
| | $7.8\times10^{-3}$ | | Raventos-Duran et al. (2010) | Q | 244 |
| | $6.2\times10^{-3}$ | | Raventos-Duran et al. (2010) | Q | 245 |
| | $6.1\times10^{-3}$ | | Zhang et al. (2010) | Q | 287, 288 |
| | $8.4\times10^{-3}$ | | Zhang et al. (2010) | Q | 287, 289 |
| | $1.8\times10^{-2}$ | | Zhang et al. (2010) | Q | 287, 290 |
| | $4.8\times10^{-3}$ | | Zhang et al. (2010) | Q | 287, 291 |
| | $9.2\times10^{-3}$ | | Hilal et al. (2008) | Q | |
| | $6.1\times10^{-3}$ | | Modarresi et al. (2007) | Q | 67 |
| | $6.8\times10^{-3}$ | | Delgado and Alderete (2002) | Q | |
| | $4.2\times10^{-3}$ | | English and Carroll (2001) | Q | 230, 231 |





Table A6.1: Chlorocarbons (C, H, Cl) (...continued)

| Substance Formula (Trivial Name) [CAS Registry Number] InChIKey | $H_s^{cp}$ (at $T^\ominus$) $\left[\dfrac{\mathrm{mol}}{\mathrm{m}^3\,\mathrm{Pa}}\right]$ | $\dfrac{\mathrm{d}\ln H_s^{cp}}{\mathrm{d}(1/T)}$ [K] | Reference | Type | Note |
|---|---|---|---|---|---|
| | $3.2\times10^{-2}$ | | Nirmalakhandan et al. (1997) | Q | |
| | $5.7\times10^{-3}$ | | Myrdal and Yalkowsky (1994) | Q | |
| | $6.1\times10^{-3}$ | | Meylan and Howard (1991) | Q | |
| pentachlorobenzene | $1.4\times10^{-2}$ | 5400 | Schwardt et al. (2021) | L | 1 |
| $C_6HCl_5$ | $1.4\times10^{-2}$ | 5200 | Shen and Wania (2005) | L | 366 |
| [608-93-5] | $1.4\times10^{-2}$ | 5600 | Shen and Wania (2005) | L | 367 |
| CEOCDNVZRAIOQZ-UHFFFAOYSA-N | $3.0\times10^{-2}$ | | McPhedran et al. (2013) | M | |
| | $5.6\times10^{-3}$ | | Lee et al. (2012) | M | |
| | $1.4\times10^{-2}$ | 5200 | ten Hulscher et al. (1992) | M | |
| | $2.0\times10^{-1}$ | | Hellmann (1987) | M | 87 |
| | $1.4\times10^{-2}$ | | Oliver (1985) | M | |
| | $1.2\times10^{-2}$ | | Mackay et al. (2006b) | V | |
| | $3.5\times10^{-2}$ | | Fogg and Sangster (2003) | V | |
| | $2.4\times10^{-2}$ | | Fogg and Sangster (2003) | V | |
| | $1.2\times10^{-2}$ | | Shiu and Mackay (1997) | V | |
| | $1.2\times10^{-2}$ | | Mackay et al. (1992a) | V | |
| | $1.2\times10^{-2}$ | | Bobra et al. (1985) | V | |
| | $1.0\times10^{-3}$ | | Mackay and Shiu (1981) | V | |
| | $7.4\times10^{-3}$ | | Keshavarz et al. (2022) | Q | |
| | $1.5\times10^{-2}$ | | Duchowicz et al. (2020) | Q | 299 |
| | $9.9\times10^{-3}$ | | Raventos-Duran et al. (2010) | Q | 271, 243 |
| | $6.2\times10^{-3}$ | | Raventos-Duran et al. (2010) | Q | 244 |
| | $7.8\times10^{-3}$ | | Raventos-Duran et al. (2010) | Q | 245 |
| | $8.2\times10^{-3}$ | | Zhang et al. (2010) | Q | 287, 288 |
| | $6.9\times10^{-3}$ | | Zhang et al. (2010) | Q | 287, 289 |
| | $1.8\times10^{-2}$ | | Zhang et al. (2010) | Q | 287, 290 |
| | $7.0\times10^{-3}$ | | Zhang et al. (2010) | Q | 287, 291 |
| | $7.2\times10^{-3}$ | | Hilal et al. (2008) | Q | |
| | $1.1\times10^{-2}$ | | Modarresi et al. (2007) | Q | 67 |
| | | 5700 | Kühne et al. (2005) | Q | |
| | $1.0\times10^{-3}$ | | Yaffe et al. (2003) | Q | 248, 249 |
| | $7.9\times10^{-3}$ | | Delgado and Alderete (2002) | Q | |
| | $9.4\times10^{-3}$ | | Myrdal and Yalkowsky (1994) | Q | |
| | $1.4\times10^{-2}$ | | Duchowicz et al. (2020) | ? | 185, 21 |
| | | 5100 | Kühne et al. (2005) | ? | |
| hexachlorobenzene | $6.5\times10^{-3}$ | 6600 | Brockbank (2013) | L | 1 |
| $C_6Cl_6$ | $1.9\times10^{-2}$ | 6000 | Shen and Wania (2005) | L | 366 |
| [118-74-1] | $1.5\times10^{-2}$ | 6400 | Shen and Wania (2005) | L | 367 |
| CKAPSXZOOQJIBF-UHFFFAOYSA-N | $3.3\times10^{-2}$ | | McPhedran et al. (2013) | M | |
| | $7.6\times10^{-3}$ | | Lee et al. (2012) | M | |
| | $3.0\times10^{-2}$ | 6900 | Jantunen and Bidleman (2006) | M | |
| | $4.2\times10^{-2}$ | | Altschuh et al. (1999) | M | |
| | $3.8\times10^{-5}$ | 570 | Hansen et al. (1993) | M | 281 |
| | $2.0\times10^{-2}$ | 5700 | ten Hulscher et al. (1992) | M | |
| | 2.6 | | Hellmann (1987) | M | 87 |
| | $2.1\times10^{-2}$ | | Oliver (1985) | M | |





Table A6.1: Chlorocarbons (C, H, Cl) (...continued)

| Substance<br>Formula<br>(Trivial Name)<br>[CAS Registry Number]<br>InChIKey | $H_s^{cp}$<br>(at $T^\ominus$)<br>$\left[\dfrac{\text{mol}}{\text{m}^3\,\text{Pa}}\right]$ | $\dfrac{\text{d}\ln H_s^{cp}}{\text{d}(1/T)}$<br><br>[K] | Reference | Type | Note |
|---|---|---|---|---|---|
| | $1.4\times10^{-2}$ | | Atlas et al. (1983) | M | 72 |
| | $7.5\times10^{-3}$ | | Atlas et al. (1982) | M | 679 |
| | $5.8\times10^{-3}$ | | Warner et al. (1980) | M | |
| | $7.6\times10^{-3}$ | | Mackay et al. (2006b) | V | |
| | $7.6\times10^{-3}$ | | Shiu and Mackay (1997) | V | |
| | $7.7\times10^{-3}$ | | Lide and Frederikse (1995) | V | |
| | $7.6\times10^{-3}$ | | Mackay et al. (1992a) | V | |
| | $7.1\times10^{-3}$ | | Ballschmiter and Wittlinger (1991) | V | |
| | $1.1\times10^{-2}$ | | Calamari et al. (1991) | V | 12 |
| | $1.4\times10^{-1}$ | | Riederer (1990) | V | |
| | $2.5\times10^{-2}$ | | McLachlan et al. (1990) | V | 373 |
| | $1.4\times10^{-1}$ | | Suntio et al. (1988) | V | 12 |
| | $7.2\times10^{-3}$ | | Bobra et al. (1985) | V | |
| | $1.6\times10^{-2}$ | | Yoshida et al. (1983) | V | |
| | $2.0\times10^{-1}$ | | Mackay and Shiu (1981) | V | |
| | $3.0\times10^{-3}$ | 3700 | Paasivirta et al. (1999) | T | |
| | $2.6\times10^{-3}$ | | Yaws (2003) | X | 237 |
| | $5.8\times10^{-3}$ | 1600 | Goldstein (1982) | X | 298 |
| | $1.0\times10^{-2}$ | | Hilal et al. (2008) | C | |
| | $1.5\times10^{-2}$ | | Suntio et al. (1988) | C | 12 |
| | $5.8\times10^{-3}$ | | Ryan et al. (1988) | C | |
| | $5.8\times10^{-3}$ | | Shen (1982) | C | |
| | $7.4\times10^{-3}$ | | Keshavarz et al. (2022) | Q | |
| | $1.4\times10^{-2}$ | | Duchowicz et al. (2020) | Q | 184 |
| | $1.1\times10^{-2}$ | | Zhang et al. (2010) | Q | 287, 288 |
| | $6.1\times10^{-3}$ | | Zhang et al. (2010) | Q | 287, 289 |
| | $1.6\times10^{-2}$ | | Zhang et al. (2010) | Q | 287, 290 |
| | $1.0\times10^{-2}$ | | Zhang et al. (2010) | Q | 287, 291 |
| | $2.3\times10^{-3}$ | | Gharagheizi et al. (2010) | Q | 246 |
| | $2.0\times10^{-2}$ | | Hilal et al. (2008) | Q | |
| | $1.4\times10^{-2}$ | | Modarresi et al. (2007) | Q | 67 |
| | | 6400 | Kühne et al. (2005) | Q | |
| | $2.1\times10^{-1}$ | | Yaffe et al. (2003) | Q | 248, 249 |
| | $6.5\times10^{-3}$ | | Delgado and Alderete (2002) | Q | |
| | $1.6\times10^{-2}$ | | Myrdal and Yalkowsky (1994) | Q | |
| | $8.6\times10^{-3}$ | | Nirmalakhandan and Speece (1988) | Q | |
| | $5.8\times10^{-3}$ | | Duchowicz et al. (2020) | ? | 185, 21 |
| | | 7200 | Kühne et al. (2005) | ? | |
| | $2.4\times10^{-5}$ | | Yaws and Yang (1992) | ? | 21 |
| (chloromethyl)-benzene<br>$C_6H_5CH_2Cl$<br>(benzylchloride)<br>[100-44-7]<br>KCXMKQUNVWSEMD-UHFFFAOYSA-N | $2.2\times10^{-2}$ | 4400 | Brockbank (2013) | L | 1 |
| | $2.0\times10^{-2}$ | 7200 | Hiatt (2013) | M | |
| | $2.8\times10^{-2}$ | | Hovorka and Dohnal (1997) | M | 12 |
| | $1.2\times10^{-2}$ | | Li and Carr (1993) | M | |
| | $2.4\times10^{-2}$ | | Duchowicz et al. (2020) | V | 186 |
| | $2.4\times10^{-2}$ | | HSDB (2015) | V | |
| | $2.9\times10^{-2}$ | | Lide and Frederikse (1995) | V | |
| | $1.6\times10^{-2}$ | | Mackay and Shiu (1981) | V | |



Table A6.1: Chlorocarbons (C, H, Cl) (...continued)

| Substance<br>Formula<br>(Trivial Name)<br>[CAS Registry Number]<br>InChIKey | $H_s^{cp}$<br>(at $T^\ominus$)<br>$\left[\dfrac{\mathrm{mol}}{\mathrm{m^3\,Pa}}\right]$ | $\dfrac{\mathrm{d}\ln H_s^{cp}}{\mathrm{d}(1/T)}$<br><br>[K] | Reference | Type | Note |
|---|---|---|---|---|---|
| | $8.6\times10^{-3}$ | | Duchowicz et al. (2020) | Q | |
| | $4.9\times10^{-2}$ | | Raventos-Duran et al. (2010) | Q | 271, 243 |
| | $3.1\times10^{-2}$ | | Raventos-Duran et al. (2010) | Q | 244 |
| | $4.9\times10^{-3}$ | | Raventos-Duran et al. (2010) | Q | 245 |
| | $3.0\times10^{-2}$ | | Hilal et al. (2008) | Q | |
| | $1.9\times10^{-2}$ | | Modarresi et al. (2007) | Q | 67 |
| | $1.6\times10^{-2}$ | | Yaffe et al. (2003) | Q | 248, 249 |
| | $2.3\times10^{-2}$ | | Katritzky et al. (1998) | Q | |
| | $1.0\times10^{-2}$ | | Abraham et al. (1990) | ? | |
| 1-chloro-2-methylbenzene<br>$C_7H_7Cl$<br>($o$-chlorotoluene)<br>[95-49-8]<br>IBSQPLPBRSHTTG-UHFFFAOYSA-N | $2.2\times10^{-3}$ | 4500 | Brockbank (2013) | L | 1 |
| | $3.2\times10^{-3}$ | 4100 | Hiatt (2013) | M | |
| | $2.4\times10^{-3}$ | 3400 | Kondoh and Nakajima (1997) | M | |
| | $2.8\times10^{-3}$ | 3500 | Leighton and Calo (1981) | M | |
| | $1.9\times10^{-2}$ | 3000 | Goldstein (1982) | X | 298 |
| | $2.8\times10^{-3}$ | | Schüürmann (2000) | C | 21 |
| | $5.3\times10^{-2}$ | | Keshavarz et al. (2022) | Q | |
| | $4.3\times10^{-3}$ | | Duchowicz et al. (2020) | Q | 299 |
| | $4.3\times10^{-3}$ | | Hilal et al. (2008) | Q | |
| | $5.4\times10^{-3}$ | | Modarresi et al. (2007) | Q | 67 |
| | | 4400 | Kühne et al. (2005) | Q | |
| | $5.7\times10^{-3}$ | | Katritzky et al. (1998) | Q | |
| | $3.1\times10^{-3}$ | | Nirmalakhandan et al. (1997) | Q | |
| | $2.8\times10^{-3}$ | | Duchowicz et al. (2020) | ? | 185, 21 |
| | | 4900 | Kühne et al. (2005) | ? | |
| | $2.8\times10^{-3}$ | | Abraham et al. (1990) | ? | |
| 1-chloro-3-methylbenzene<br>$C_7H_7Cl$<br>($m$-chlorotoluene)<br>[108-41-8]<br>OSOUNOBYRMOXQQ-UHFFFAOYSA-N | $6.1\times10^{-4}$ | | Duchowicz et al. (2020) | V | 186 |
| | $6.2\times10^{-4}$ | | Schüürmann (2000) | V | |
| | $4.3\times10^{-3}$ | | Duchowicz et al. (2020) | Q | |
| | $3.8\times10^{-3}$ | | Hilal et al. (2008) | Q | |
| | $6.4\times10^{-3}$ | | Modarresi et al. (2007) | Q | 67 |
| | | 4400 | Kühne et al. (2005) | Q | |
| | $6.1\times10^{-4}$ | | Yaffe et al. (2003) | Q | 248, 249 |
| | $4.8\times10^{-3}$ | | Katritzky et al. (1998) | Q | |
| | | 4800 | Kühne et al. (2005) | ? | |
| 1-chloro-4-methylbenzene<br>$C_7H_7Cl$<br>($p$-chlorotoluene)<br>[106-43-4]<br>NPDACUSDTOMAMK-UHFFFAOYSA-N | $2.7\times10^{-3}$ | 4900 | Brockbank (2013) | L | 1 |
| | $4.1\times10^{-3}$ | 4200 | Hiatt (2013) | M | |
| | $2.9\times10^{-3}$ | 3900 | Kondoh and Nakajima (1997) | M | |
| | $2.3\times10^{-3}$ | | Duchowicz et al. (2020) | V | 186 |
| | $2.2\times10^{-3}$ | | HSDB (2015) | V | |
| | $3.1\times10^{-3}$ | | Yaws (2003) | X | 237, 12 |
| | $4.3\times10^{-3}$ | | Duchowicz et al. (2020) | Q | |
| | $8.0\times10^{-3}$ | | Gharagheizi et al. (2012) | Q | |
| | $3.3\times10^{-3}$ | | Gharagheizi et al. (2010) | Q | 246 |
| | $4.0\times10^{-3}$ | | Hilal et al. (2008) | Q | |
| | $6.8\times10^{-3}$ | | Modarresi et al. (2007) | Q | 67 |
| | | 4400 | Kühne et al. (2005) | Q | |



Table A6.1: Chlorocarbons (C, H, Cl) (...continued)

| Substance<br>Formula<br>(Trivial Name)<br>[CAS Registry Number]<br>InChIKey | $H_s^{cp}$<br>(at $T^{\ominus}$)<br>$\left[\dfrac{\text{mol}}{\text{m}^3\,\text{Pa}}\right]$ | $\dfrac{\text{d}\ln H_s^{cp}}{\text{d}(1/T)}$<br><br>[K] | Reference | Type | Note |
|---|---|---|---|---|---|
| | $5.1\times10^{-3}$ | | Yao et al. (2002) | Q | 229 |
| | $4.8\times10^{-3}$ | | Katritzky et al. (1998) | Q | |
| | | 4300 | Kühne et al. (2005) | ? | |
| | $2.3\times10^{-3}$ | | Yaws (1999) | ? | 21, 12 |
| (dichloromethyl)-benzene | $2.5\times10^{-2}$ | | Duchowicz et al. (2020) | V | 186 |
| $C_7H_6Cl_2$ | $1.8\times10^{-2}$ | | Duchowicz et al. (2020) | Q | |
| [98-87-3] | $1.3\times10^{-2}$ | | Zhang et al. (2010) | Q | 287, 288 |
| CAHQGWAXKLQREW-UHFFFAOYSA-N | $3.4\times10^{-2}$ | | Zhang et al. (2010) | Q | 287, 289 |
| | $1.1\times10^{-1}$ | | Zhang et al. (2010) | Q | 287, 290 |
| | $1.0\times10^{-2}$ | | Zhang et al. (2010) | Q | 287, 291 |
| | $3.9\times10^{-2}$ | | Hilal et al. (2008) | Q | |
| | $2.0\times10^{-2}$ | | Modarresi et al. (2007) | Q | 67 |
| 1,2-dichloro-4-methylbenzene | $3.8\times10^{-3}$ | | Duchowicz et al. (2020) | V | 186 |
| $C_7H_6Cl_2$ | $7.9\times10^{-3}$ | | Duchowicz et al. (2020) | Q | |
| (3,4-dichlorotoluene) | $7.9\times10^{-3}$ | | Hilal et al. (2008) | Q | |
| [95-75-0] | $4.1\times10^{-3}$ | | Modarresi et al. (2007) | Q | 67 |
| WYUIWKFIFOJVKW-UHFFFAOYSA-N | $4.1\times10^{-3}$ | | Yaffe et al. (2003) | Q | 248, 249 |
| | $9.0\times10^{-3}$ | | Katritzky et al. (1998) | Q | |
| 1,3-dichloro-2-methylbenzene | $2.3\times10^{-3}$ | | HSDB (2015) | Q | 99 |
| $C_7H_6Cl_2$ | $3.1\times10^{-3}$ | | Zhang et al. (2010) | Q | 287, 288 |
| (2,6-dichlorotoluene) | $8.6\times10^{-3}$ | | Zhang et al. (2010) | Q | 287, 289 |
| [118-69-4] | $4.2\times10^{-3}$ | | Zhang et al. (2010) | Q | 287, 290 |
| DMEDNTFWIHCBRK-UHFFFAOYSA-N | $1.8\times10^{-3}$ | | Zhang et al. (2010) | Q | 287, 291 |
| 2,4-dichloro-1-methylbenzene | $2.7\times10^{-3}$ | 5000 | Schwardt et al. (2021) | L | 1 |
| $C_7H_6Cl_2$ | $2.7\times10^{-3}$ | 5000 | Brockbank (2013) | L | 1 |
| (2,4-dichlorotoluene) | $2.7\times10^{-3}$ | 4900 | Brockbank et al. (2013) | M | |
| [95-73-8] | $2.3\times10^{-3}$ | | HSDB (2015) | Q | 99 |
| FUNUTBJJKQIVSY-UHFFFAOYSA-N | $3.1\times10^{-3}$ | | Zhang et al. (2010) | Q | 287, 288 |
| | $5.4\times10^{-3}$ | | Zhang et al. (2010) | Q | 287, 289 |
| | $6.7\times10^{-3}$ | | Zhang et al. (2010) | Q | 287, 290 |
| | $1.8\times10^{-3}$ | | Zhang et al. (2010) | Q | 287, 291 |
| | | 4400 | Kühne et al. (2005) | Q | |
| | | 5500 | Kühne et al. (2005) | ? | |
| 2,3,6-trichloromethylbenzene | $6.6\times10^{-3}$ | | Oliver (1985) | M | |
| $C_7H_5Cl_3$ | $1.4\times10^{-2}$ | | Hilal et al. (2008) | Q | |
| (2,3,6-trichlorotoluene) | $6.0\times10^{-3}$ | | Modarresi et al. (2007) | Q | 67 |
| [2077-46-5] | $4.1\times10^{-3}$ | | Meylan and Howard (1991) | Q | |
| UZYYBZNZSSNYSA-UHFFFAOYSA-N | | | | | |
| 2,4,5-trichloromethylbenzene | $6.6\times10^{-3}$ | | Oliver (1985) | M | |
| $C_7H_5Cl_3$ | $1.2\times10^{-2}$ | | Hilal et al. (2008) | Q | |
| (2,4,5-trichlorotoluene) | $6.2\times10^{-3}$ | | Modarresi et al. (2007) | Q | 67 |
| [6639-30-1] | $4.1\times10^{-3}$ | | Meylan and Howard (1991) | Q | |
| ZCXHZKNWIYVQNC-UHFFFAOYSA-N | | | | | |



Table A6.1: Chlorocarbons (C, H, Cl) (... continued)

| Substance<br>Formula<br>(Trivial Name)<br>[CAS Registry Number]<br>InChIKey | $H_s^{cp}$<br>(at $T^\ominus$)<br>$\left[\dfrac{\mathrm{mol}}{\mathrm{m^3\,Pa}}\right]$ | $\dfrac{\mathrm{d}\ln H_s^{cp}}{\mathrm{d}(1/T)}$<br><br>[K] | Reference | Type | Note |
|---|---|---|---|---|---|
| 2,4,6-trichloromethylbenzene<br>$C_7H_5Cl_3$<br>(2,4,6-trichlorotoluene)<br>[23749-65-7]<br>RCTKUIOMKBEGTG-UHFFFAOYSA-N | $6.5\times10^{-3}$ | | Ebert et al. (2023) | ? | 365 |
| pentachloromethylbenzene<br>$C_7H_3Cl_5$<br>(2,3,4,5,6-pentachlorotoluene)<br>[877-11-2]<br>AVSIMRGRHWKCAY-UHFFFAOYSA-N | $1.3\times10^{-2}$<br>$1.6\times10^{-2}$<br>$1.7\times10^{-2}$<br>$7.4\times10^{-3}$ | | Oliver (1985)<br>Hilal et al. (2008)<br>Modarresi et al. (2007)<br>Meylan and Howard (1991) | M<br>Q<br>Q<br>Q | <br><br>67<br> |
| 1-chloro-2-(chloromethyl)benzene<br>$C_7H_6Cl_2$<br>[611-19-8]<br>BASMANVIUSSIIM-UHFFFAOYSA-N | $6.4\times10^{-3}$<br>$7.2\times10^{-2}$<br>$2.1\times10^{-1}$<br>$6.1\times10^{-3}$ | | Zhang et al. (2010)<br>Zhang et al. (2010)<br>Zhang et al. (2010)<br>Zhang et al. (2010) | Q<br>Q<br>Q<br>Q | 287, 288<br>287, 289<br>287, 290<br>287, 291 |
| 1-chloro-4-(chloromethyl)benzene<br>$C_7H_6Cl_2$<br>[104-83-6]<br>JQZAEUFPPSRDOP-UHFFFAOYSA-N | $2.9\times10^{-2}$<br>$6.4\times10^{-3}$<br>$7.5\times10^{-2}$<br>$8.2\times10^{-2}$<br>$6.1\times10^{-3}$ | | HSDB (2015)<br>Zhang et al. (2010)<br>Zhang et al. (2010)<br>Zhang et al. (2010)<br>Zhang et al. (2010) | Q<br>Q<br>Q<br>Q<br>Q | 99<br>287, 288<br>287, 289<br>287, 290<br>287, 291 |
| chloro(dichloromethyl)benzene<br>$C_7H_5Cl_3$<br>[88-66-4]<br>BXSVYGKOUULJCL-UHFFFAOYSA-N | $1.8\times10^{-2}$<br>$7.3\times10^{-2}$<br>$5.4\times10^{-2}$<br>$1.3\times10^{-2}$ | | Zhang et al. (2010)<br>Zhang et al. (2010)<br>Zhang et al. (2010)<br>Zhang et al. (2010) | Q<br>Q<br>Q<br>Q | 287, 288<br>287, 289<br>287, 290<br>287, 291 |
| (trichloromethyl)-benzene<br>$C_7H_5Cl_3$<br>[98-07-7]<br>XEMRAKSQROQPBR-UHFFFAOYSA-N | $3.8\times10^{-2}$<br>$3.8\times10^{-2}$<br>$7.2\times10^{-3}$<br>$2.0\times10^{-2}$<br>$4.7\times10^{-3}$ | | HSDB (2015)<br>Zhang et al. (2010)<br>Zhang et al. (2010)<br>Zhang et al. (2010)<br>Zhang et al. (2010) | Q<br>Q<br>Q<br>Q<br>Q | 99<br>287, 288<br>287, 289<br>287, 290<br>287, 291 |
| 1-chloro-4-<br>(trichloromethyl)benzene<br>$C_7H_4Cl_4$<br>[5216-25-1]<br>LVZPKYYPPLUECL-UHFFFAOYSA-N | $5.1\times10^{-2}$<br><br>$1.8\times10^{-2}$<br>$3.4\times10^{-2}$<br>$6.9\times10^{-3}$ | | Zhang et al. (2010)<br><br>Zhang et al. (2010)<br>Zhang et al. (2010)<br>Zhang et al. (2010) | Q<br><br>Q<br>Q<br>Q | 287, 288<br><br>287, 289<br>287, 290<br>287, 291 |
| 1-chloro-3-ethenylbenzene<br>$C_8H_7Cl$<br>[2039-85-2]<br>BOVQCIDBZXNFEJ-UHFFFAOYSA-N | $4.7\times10^{-3}$ | | HSDB (2015) | Q | 99 |
| 1-chloro-4-ethenylbenzene<br>$C_8H_7Cl$<br>[1073-67-2]<br>KTZVZZJJVJQZHV-UHFFFAOYSA-N | $4.7\times10^{-3}$ | | HSDB (2015) | Q | 99 |



Table A6.1: Chlorocarbons (C, H, Cl) (. . . continued)

| Substance Formula (Trivial Name) [CAS Registry Number] InChIKey | $H_s^{cp}$ (at $T^{\ominus}$) $\left[\dfrac{\mathrm{mol}}{\mathrm{m^3\,Pa}}\right]$ | $\dfrac{\mathrm{d}\ln H_s^{cp}}{\mathrm{d}(1/T)}$ [K] | Reference | Type | Note |
|---|---|---|---|---|---|
| 1,4-dichloro-2,5-dimethylbenzene $C_8H_8Cl_2$ [1124-05-6] UTGSRNVBAFCOEU-UHFFFAOYSA-N | $2.7\times10^{-3}$ $1.2\times10^{-2}$ $4.6\times10^{-3}$ $2.3\times10^{-3}$ | | Zhang et al. (2010) Zhang et al. (2010) Zhang et al. (2010) Zhang et al. (2010) | Q Q Q Q | 287, 288 287, 289 287, 290 287, 291 |
| 1,4-bis(trichloromethyl)benzene $C_8H_4Cl_6$ [68-36-0] OTEKOJQFKOIXMU-UHFFFAOYSA-N | $7.9\times10^{-1}$ $3.7\times10^{-2}$ $1.1\times10^{-1}$ $5.8\times10^{-3}$ | | Zhang et al. (2010) Zhang et al. (2010) Zhang et al. (2010) Zhang et al. (2010) | Q Q Q Q | 287, 288 287, 289 287, 290 287, 291 |
| $\alpha,\alpha$-dichloro-$o$-xylene $C_8H_8Cl_2$ [612-12-4] FMGGHNGKHRCJLL-UHFFFAOYSA-N | $1.0\times10^{-1}$ | 11000 | Hiatt (2013) | M | |
| 2-chlorostyrene $C_8H_7Cl$ [2039-87-4] ISRGONDNXBCDBM-UHFFFAOYSA-N | $4.7\times10^{-3}$ $6.2\times10^{-3}$ | | HSDB (2015) Hilal et al. (2008) | Q Q | 99 |
| octachlorostyrene $C_8Cl_8$ [29082-74-4] RUYUCCQRWINUHE-UHFFFAOYSA-N | $7.6\times10^{-2}$ $4.3\times10^{-2}$ $1.6\times10^{-2}$ $4.3\times10^{-2}$ | | Oliver (1985) HSDB (2015) Hilal et al. (2008) Meylan and Howard (1991) | M Q Q Q | 99 |
| heptachlor $C_{10}H_5Cl_7$ [76-44-8] FRCCEHPWNOQAEU-UHFFFAOYSA-N | $3.3\times10^{-2}$ $2.6\times10^{-2}$ $1.9\times10^{-2}$ $3.4\times10^{-2}$ $6.7\times10^{-3}$ $2.8\times10^{-3}$ $8.9\times10^{-3}$ $8.8\times10^{-5}$ $4.3\times10^{-3}$ $6.7\times10^{-3}$ $6.5\times10^{-3}$ $6.7\times10^{-3}$ $1.3\times10^{-1}$ $2.9\times10^{-2}$ $2.0\times10^{-4}$ $2.4\times10^{-2}$ $5.8\times10^{-1}$ $5.6\times10^{-2}$ $3.4\times10^{-2}$ $2.8\times10^{-3}$ | 4300 | Shen and Wania (2005) Shen and Wania (2005) Cetin et al. (2006) Altschuh et al. (1999) Warner et al. (1980) Mackay et al. (2006d) Suntio et al. (1988) Barcelo and Hennion (1997) McCarty (1980) Meylan and Howard (1991) Ryan et al. (1988) Shen (1982) Keshavarz et al. (2022) Duchowicz et al. (2020) Goodarzi et al. (2010) Hilal et al. (2008) Modarresi et al. (2007) Meylan and Howard (1991) Duchowicz et al. (2020) MacBean (2012a) | L L M M M V V X X C C C Q Q Q Q Q Q ? ? | 366 367 12 567 368 568 67 185, 21 |





Table A6.1: Chlorocarbons (C, H, Cl) (. . . continued)

| Substance<br>Formula<br>(Trivial Name)<br>[CAS Registry Number]<br>InChIKey | $H_s^{cp}$<br>(at $T^\ominus$)<br>$\left[\dfrac{\mathrm{mol}}{\mathrm{m^3\,Pa}}\right]$ | $\dfrac{\mathrm{d\ln} H_s^{cp}}{\mathrm{d}(1/T)}$<br><br>[K] | Reference | Type | Note |
|---|---|---|---|---|---|
| 1-chloro-2-methyl-2-phenylpropane | $2.0\times10^{-3}$ | | Zhang et al. (2010) | Q | 287, 288 |
| $C_{10}H_{13}Cl$ | $9.5\times10^{-3}$ | | Zhang et al. (2010) | Q | 287, 289 |
| [515-40-2] | $1.6\times10^{-2}$ | | Zhang et al. (2010) | Q | 287, 290 |
| DNXXUUPUQXSUFH-UHFFFAOYSA-N | $1.6\times10^{-3}$ | | Zhang et al. (2010) | Q | 287, 291 |
| 1,3-dichloro-5-[(2$S$)-2,4,4,4-tetrachlorobutan-2-yl]benzene | $8.4\times10^{-2}$ | | Zhang et al. (2010) | Q | 287, 288 |
| $C_{10}H_8Cl_6$ | $6.1\times10^{-2}$ | | Zhang et al. (2010) | Q | 287, 289 |
| [73588-42-8] | 1.5 | | Zhang et al. (2010) | Q | 287, 290 |
| DELZPCKTBPIBEE-VIFPVBQESA-N | $2.9\times10^{-3}$ | | Zhang et al. (2010) | Q | 287, 291 |
| 1,2,3,4,5,6,7,8,8-nonachloro-2,3,3a,4,7,7a-hexahydro-4,7-methano-1H-indene<br>$C_{10}H_5Cl_9$<br>(nonachlor)<br>[3734-49-4]<br>OCHOKXCPKDPNQU-UHFFFAOYSA-N | $3.9\times10^{-1}$ | | HSDB (2015) | Q | 99 |
| 1,1-dichloro-2,2-bis-(4-chlorophenyl)-ethane | 1.5 | | Shen and Wania (2005) | L | 366 |
| $C_{14}H_{10}Cl_4$ | 2.0 | | Shen and Wania (2005) | L | 367 |
| (p,p'-DDD) | $7.6\times10^{-1}$ | | Chao et al. (2017) | M | |
| [72-54-8] | $9.1\times10^{-1}$ | 5100 | Cetin et al. (2006) | M | |
| AHJKRLASYNVKDZ-UHFFFAOYSA-N | 1.5 | | Altschuh et al. (1999) | M | |
| | | | Mackay et al. (2006d) | V | 558 |
| | $1.1\times10^{-1}$ | | Ballschmiter and Wittlinger (1991) | V | |
| | 1.6 | | Suntio et al. (1988) | V | 12 |
| | $4.6\times10^{-1}$ | | Yoshida et al. (1983) | V | |
| | $2.9\times10^{-2}$ | 7300 | Paasivirta et al. (1999) | T | |
| | $8.1\times10^{-4}$ | | Ryan et al. (1988) | C | |
| | $4.3\times10^{-1}$ | | Keshavarz et al. (2022) | Q | |
| | $9.5\times10^{-2}$ | | Duchowicz et al. (2020) | Q | 184 |
| | 2.1 | | Hilal et al. (2008) | Q | |
| | $3.1\times10^{-1}$ | | Modarresi et al. (2007) | Q | 67 |
| | 1.5 | | Duchowicz et al. (2020) | ? | 185, 21 |
| mitotane | 1.2 | | Duchowicz et al. (2020) | V | 186 |
| $C_{14}H_{10}Cl_4$ | 1.2 | | HSDB (2015) | V | |
| (o,p'-DDD) | 1.6 | | Suntio et al. (1988) | V | 12 |
| [53-19-0] | $5.6\times10^2$ | | Suntio et al. (1988) | C | 715 |
| JWBOIMRXGHLCPP-UHFFFAOYSA-N | $9.5\times10^{-2}$ | | Duchowicz et al. (2020) | Q | |
| | $2.3\times10^{-1}$ | | Zhang et al. (2010) | Q | 287, 288 |
| | 1.6 | | Zhang et al. (2010) | Q | 287, 289 |
| | $1.1\times10^1$ | | Zhang et al. (2010) | Q | 287, 290 |
| | $3.9\times10^{-1}$ | | Zhang et al. (2010) | Q | 287, 291 |



Table A6.1: Chlorocarbons (C, H, Cl) (. . . continued)

| Substance<br>Formula<br>(Trivial Name)<br>[CAS Registry Number]<br>InChIKey | $H_s^{cp}$<br>(at $T^{\ominus}$)<br>$\left[\dfrac{\text{mol}}{\text{m}^3\,\text{Pa}}\right]$ | $\dfrac{\text{d}\ln H_s^{cp}}{\text{d}(1/T)}$<br><br>[K] | Reference | Type | Note |
|---|---|---|---|---|---|
| 1,1-dichloro-2,2-bis-(4-<br>chlorophenyl)-ethene | $2.4\times10^{-1}$ | | Shen and Wania (2005) | L | 366 |
| $C_{14}H_8Cl_4$ | $2.4\times10^{-1}$ | | Shen and Wania (2005) | L | 367 |
| (p,p'-DDE) | $2.9\times10^{-2}$ | 4700 | Jantunen and Bidleman (2006) | M | |
| [72-55-9] | $1.6\times10^{-1}$ | 7700 | Cetin et al. (2006) | M | |
| UCNVFOCBFJOQAL-UHFFFAOYSA-N | $2.4\times10^{-1}$ | | Altschuh et al. (1999) | M | |
| | $8.1\times10^{-3}$ | | Atlas et al. (1982) | M | 679 |
| | | | Mackay et al. (2006d) | V | 558 |
| | $2.9\times10^{-2}$ | | Ballschmiter and Wittlinger (1991) | V | |
| | $1.6\times10^{-1}$ | | Calamari et al. (1991) | V | 12 |
| | $7.6\times10^{-1}$ | | McLachlan et al. (1990) | V | 373 |
| | $1.3\times10^{-1}$ | | Suntio et al. (1988) | V | 12 |
| | $5.1\times10^{-2}$ | | Yoshida et al. (1983) | V | |
| | $1.2\times10^{-1}$ | | Addison et al. (1983) | V | |
| | $2.6\times10^{-2}$ | 7600 | Paasivirta et al. (1999) | T | |
| | $4.5\times10^{-1}$ | | Suntio et al. (1988) | C | 681 |
| | $4.5\times10^{-1}$ | | Ryan et al. (1988) | C | |
| | $8.1\times10^{-2}$ | | Keshavarz et al. (2022) | Q | |
| | $4.2\times10^{-2}$ | | Duchowicz et al. (2020) | Q | |
| | $1.8\times10^{-1}$ | | Hilal et al. (2008) | Q | |
| | $2.1\times10^{-1}$ | | Modarresi et al. (2007) | Q | 67 |
| | $2.4\times10^{-1}$ | | Duchowicz et al. (2020) | ? | 185, 21 |
| o,p'-DDE | $5.3\times10^{-1}$ | | Duchowicz et al. (2020) | V | 186 |
| $C_{14}H_8Cl_4$ | $3.9\times10^{-1}$ | | Mackay et al. (2006d) | V | |
| [3424-82-6] | $3.9\times10^{-1}$ | | Suntio et al. (1988) | V | 12 |
| ZDYJWDIWLRZXDB-UHFFFAOYSA-N | $1.4\times10^{-1}$ | | Suntio et al. (1988) | C | 12 |
| | $4.2\times10^{-2}$ | | Duchowicz et al. (2020) | Q | |
| 1,1,1-trichloro-2-(2-chlorophenyl)-<br>2-(4-chlorophenyl)ethane | 1.3 | | Duchowicz et al. (2020) | V | 186 |
| $C_{14}H_9Cl_5$ | 2.9 | | Mackay et al. (2006d) | V | |
| (o,p'-DDT) | $1.9\times10^{-2}$ | | Calamari et al. (1991) | V | 12 |
| [789-02-6] | $4.4\times10^{-2}$ | | Duchowicz et al. (2020) | Q | |
| CVUGPAFCQJIYDT-UHFFFAOYSA-N | | | | | |
| 1,1,1-trichloro-2,2-bis-(4-<br>chlorophenyl)-ethane | $9.1\times10^{-1}$ | | Shen and Wania (2005) | L | 366 |
| $C_{14}H_9Cl_5$ | $9.1\times10^{-1}$ | | Shen and Wania (2005) | L | 367 |
| (DDT; p,p'-DDT) | $1.9\times10^{-1}$ | | Mackay and Shiu (1981) | L | |
| [50-29-3] | $9.0\times10^{-1}$ | 7500 | Cetin et al. (2006) | M | |
| YVGGHNCTFXOJCH-UHFFFAOYSA-N | 1.2 | | Altschuh et al. (1999) | M | |
| | $7.7\times10^{-1}$ | | Fendinger et al. (1989) | M | 72 |
| | 1.2 | | Fendinger et al. (1989) | M | 645 |
| | | | Mackay et al. (2006d) | V | 558 |
| | $1.7\times10^{-1}$ | | Ballschmiter and Wittlinger (1991) | V | |
| | $3.4\times10^{-1}$ | | Calamari et al. (1991) | V | 12 |
| | $4.2\times10^{-1}$ | | Suntio et al. (1988) | V | 12 |
| | $6.1\times10^{-1}$ | | Caron et al. (1985) | V | |



Table A6.1: Chlorocarbons (C, H, Cl) (...continued)

| Substance Formula (Trivial Name) [CAS Registry Number] InChIKey | $H_s^{cp}$ (at $T^{\ominus}$) $\left[\dfrac{\text{mol}}{\text{m}^3\,\text{Pa}}\right]$ | $\dfrac{\text{d}\ln H_s^{cp}}{\text{d}(1/T)}$ [K] | Reference | Type | Note |
|---|---|---|---|---|---|
| | $3.7\times10^{-1}$ | | Yoshida et al. (1983) | V | |
| | $1.3\times10^{-1}$ | | Burkhard and Guth (1981) | V | |
| | $2.5\times10^{-1}$ | | Mackay and Leinonen (1975) | V | |
| | $1.9\times10^{-2}$ | 7800 | Paasivirta et al. (1999) | T | |
| | $4.2\times10^{-3}$ | | Barcelo and Hennion (1997) | X | 567 |
| | $1.7\times10^{-1}$ | | Suntio et al. (1988) | C | 681 |
| | $2.0\times10^{-1}$ | | Ryan et al. (1988) | C | |
| | $4.3\times10^{-1}$ | | Keshavarz et al. (2022) | Q | |
| | $4.4\times10^{-2}$ | | Duchowicz et al. (2020) | Q | 299 |
| | $6.4\times10^{-1}$ | | Zhang et al. (2010) | Q | 287, 288 |
| | $6.2\times10^{-1}$ | | Zhang et al. (2010) | Q | 287, 289 |
| | $1.6\times10^{1}$ | | Zhang et al. (2010) | Q | 287, 290 |
| | $2.0\times10^{-1}$ | | Zhang et al. (2010) | Q | 287, 291 |
| | $7.7\times10^{-3}$ | | Goodarzi et al. (2010) | Q | 568 |
| | $6.7\times10^{-1}$ | | Hilal et al. (2008) | Q | |
| | $3.5\times10^{-1}$ | | Modarresi et al. (2007) | Q | 67 |
| | 1.2 | | Duchowicz et al. (2020) | ? | 185, 21 |
| | $2.8\times10^{-1}$ | | Brimblecombe (1986) | ? | 80 |
| aldrin $C_{12}H_8Cl_6$ [309-00-2] QBYJBZPUGVGKQQ-SJJAEHHWSA-N | $6.7\times10^{-2}$ | | Shen and Wania (2005) | L | 366 |
| | $4.3\times10^{-2}$ | | Shen and Wania (2005) | L | 367 |
| | $3.6\times10^{-1}$ | | Mackay and Shiu (1981) | L | |
| | $2.6\times10^{-1}$ | | Chao et al. (2017) | M | |
| | $2.2\times10^{-2}$ | 3900 | Cetin et al. (2006) | M | |
| | $2.2\times10^{-1}$ | | Altschuh et al. (1999) | M | |
| | $2.0\times10^{-2}$ | | Warner et al. (1980) | M | |
| | $1.1\times10^{-2}$ | | Mackay et al. (2006d) | V | |
| | $1.1\times10^{-2}$ | | Suntio et al. (1988) | V | 12 |
| | $6.9\times10^{-1}$ | | Mackay and Leinonen (1975) | V | |
| | $1.1\times10^{-4}$ | | Barcelo and Hennion (1997) | X | 567 |
| | $2.0\times10^{-2}$ | | Hilal et al. (2008) | C | |
| | $2.0\times10^{-2}$ | | Meylan and Howard (1991) | C | |
| | $7.0\times10^{-1}$ | | Suntio et al. (1988) | C | 12 |
| | $6.1\times10^{-1}$ | | Suntio et al. (1988) | C | |
| | $2.6\times10^{-2}$ | | Suntio et al. (1988) | C | 681 |
| | $2.0\times10^{-2}$ | | Suntio et al. (1988) | C | 12 |
| | $8.2\times10^{-1}$ | | Ryan et al. (1988) | C | |
| | $2.0\times10^{-2}$ | | Shen (1982) | C | |
| | $2.4\times10^{-1}$ | | Keshavarz et al. (2022) | Q | |
| | $2.3\times10^{-2}$ | | Duchowicz et al. (2020) | Q | 184 |
| | $6.4\times10^{-4}$ | | Goodarzi et al. (2010) | Q | 568, 569 |
| | $8.6\times10^{-3}$ | | Hilal et al. (2008) | Q | |
| | $1.6\times10^{-1}$ | | Modarresi et al. (2007) | Q | 67 |
| | $2.6\times10^{-2}$ | | Meylan and Howard (1991) | Q | |
| | $2.2\times10^{-1}$ | | Duchowicz et al. (2020) | ? | 185, 21 |
| | $8.4\times10^{-1}$ | | Brimblecombe (1986) | ? | 80 |





Table A6.1: Chlorocarbons (C, H, Cl) (... continued)

| Substance<br>Formula<br>(Trivial Name)<br>[CAS Registry Number]<br><small>InChIKey</small> | $H_s^{cp}$<br>(at $T^{\ominus}$)<br>$\left[\dfrac{\mathrm{mol}}{\mathrm{m}^3\,\mathrm{Pa}}\right]$ | $\dfrac{\mathrm{d}\ln H_s^{cp}}{\mathrm{d}(1/T)}$<br><br>[K] | Reference | Type | Note |
|---|---|---|---|---|---|
| isodrin<br>$C_{12}H_8Cl_6$<br>[465-73-6]<br><small>QBYJBZPUGVGKQQ-DIFDVCDBSA-N</small> | $2.5\times10^{-2}$ | | HSDB (2015) | Q | 99 |
| 1,1'-(2,2-dichloroethylidene)bis[4-ethylbenzene<br>$C_{18}H_{20}Cl_2$<br>(perthane)<br>[72-56-0]<br><small>QFMDFTQOJHFVNR-UHFFFAOYSA-N</small> | $5.8\times10^{-2}$ | | HSDB (2015) | Q | 99 |



### A6.2 Polychlorinated naphthalenes (PCNs)

Table A6.2: Polychlorinated naphthalenes (PCNs)

| Substance Formula (Trivial Name) [CAS Registry Number] InChIKey | $H_s^{cp}$ (at $T^{\ominus}$) $\left[\dfrac{\mathrm{mol}}{\mathrm{m}^3\,\mathrm{Pa}}\right]$ | $\dfrac{\mathrm{d}\ln H_s^{cp}}{\mathrm{d}(1/T)}$ [K] | Reference | Type | Note |
|---|---|---|---|---|---|
| 1-chloronaphthalene | $2.8\times10^{-2}$ | | Shiu and Mackay (1997) | M | |
| $C_{10}H_7Cl$ | $2.8\times10^{-3}$ | | Mackay and Shiu (1981) | M | |
| (PCN-1) | $5.0\times10^{-2}$ | | Yaws (2003) | X | 237 |
| [90-13-1] | $4.7\times10^{-2}$ | | Yaws et al. (2005) | X | 446 |
| JTPNRXUCIXHOKM-UHFFFAOYSA-N | $2.4\times10^{-2}$ | | Keshavarz et al. (2022) | Q | |
| | $3.3\times10^{-2}$ | | Duchowicz et al. (2020) | Q | |
| | $1.1\times10^{-2}$ | | Li et al. (2014) | Q | 241 |
| | $3.1\times10^{-1}$ | | Gharagheizi et al. (2012) | Q | |
| | $5.3\times10^{-2}$ | | Gharagheizi et al. (2010) | Q | 246 |
| | $5.7\times10^{-2}$ | | Hilal et al. (2008) | Q | |
| | $7.3\times10^{-2}$ | | Modarresi et al. (2007) | Q | 67 |
| | $3.1\times10^{-2}$ | | Yaffe et al. (2003) | Q | 248, 272 |
| | $1.1\times10^{-2}$ | | Katritzky et al. (1998) | Q | |
| | $6.5\times10^{-2}$ | | Nirmalakhandan and Speece (1988) | Q | |
| | $2.8\times10^{-2}$ | | Duchowicz et al. (2020) | ? | 185, 21 |
| | $5.1\times10^{-2}$ | | Yaws (1999) | ? | 21 |
| 2-chloronaphthalene | $3.0\times10^{-2}$ | | Shiu and Mackay (1997) | M | |
| $C_{10}H_7Cl$ | $3.1\times10^{-2}$ | | Mackay and Shiu (1981) | M | |
| (PCN-2) | $1.5\times10^{-2}$ | | Hwang et al. (1992) | V | |
| [91-58-7] | $1.6\times10^{-2}$ | 3800 | Goldstein (1982) | X | 298 |
| CGYGETOMCSJHJU-UHFFFAOYSA-N | $3.1\times10^{-2}$ | | Ryan et al. (1988) | C | |
| | $2.4\times10^{-2}$ | | Keshavarz et al. (2022) | Q | |
| | $3.3\times10^{-2}$ | | Duchowicz et al. (2020) | Q | 184 |
| | $6.0\times10^{-2}$ | | Hilal et al. (2008) | Q | |
| | $1.0\times10^{-1}$ | | Modarresi et al. (2007) | Q | 67 |
| | $3.1\times10^{-2}$ | | Yaffe et al. (2003) | Q | 248, 249 |
| | $6.5\times10^{-2}$ | | Nirmalakhandan and Speece (1988) | Q | |
| | $3.1\times10^{-2}$ | | Duchowicz et al. (2020) | ? | 185, 21 |
| 1,2-dichloronaphthalene | $1.2\times10^{-1}$ | | Ebert et al. (2023) | ? | 716 |
| $C_{10}H_6Cl_2$ | | | | | |
| (PCN-3) | | | | | |
| [2050-69-3] | | | | | |
| MOXLHAPKZWTHEX-UHFFFAOYSA-N | | | | | |
| 1,3-dichloronaphthalene | $3.4\times10^{-2}$ | | Zhang et al. (2010) | Q | 287, 288 |
| $C_{10}H_6Cl_2$ | $7.0\times10^{-2}$ | | Zhang et al. (2010) | Q | 287, 289 |
| (PCN-4) | $4.2\times10^{-2}$ | | Zhang et al. (2010) | Q | 287, 290 |
| [2198-75-6] | $4.7\times10^{-2}$ | | Zhang et al. (2010) | Q | 287, 291 |
| AMCBMCWLCDERHY-UHFFFAOYSA-N | | | | | |



Table A6.2: Polychlorinated naphthalenes (PCNs) (...continued)

| Substance Formula (Trivial Name) [CAS Registry Number] InChIKey | $H_s^{cp}$ (at $T^{\ominus}$) $\left[\dfrac{\mathrm{mol}}{\mathrm{m^3\,Pa}}\right]$ | $\dfrac{\mathrm{d}\ln H_s^{cp}}{\mathrm{d}(1/T)}$ [K] | Reference | Type | Note |
|---|---|---|---|---|---|
| 1,4-dichloronaphthalene $C_{10}H_6Cl_2$ (PCN-5) [1825-31-6] JDPKCYMVSKDOGS-UHFFFAOYSA-N | $2.4\times10^{-2}$ $3.2\times10^{-2}$ | | Duchowicz et al. (2020) Duchowicz et al. (2020) | V Q | 186 |
| 1,5-dichloronaphthalene $C_{10}H_6Cl_2$ (PCN-6) [1825-30-5] ZBQZXTBAGBTUAD-UHFFFAOYSA-N | $4.0\times10^{-2}$ | | Ebert et al. (2023) | ? | 716 |
| 2,7-dichloronaphthalene $C_{10}H_6Cl_2$ (PCN-12) [2198-77-8] DWBQZSYTSNYEEJ-UHFFFAOYSA-N | $3.1\times10^{-2}$ | | Ebert et al. (2023) | ? | 717 |
| 1,2,4-trichloronaphthalene $C_{10}H_5Cl_3$ (PCN-14) [50402-51-2] MRJBOPVWPORBKZ-UHFFFAOYSA-N | $8.3\times10^{-2}$ | | Ebert et al. (2023) | ? | 365 |
| 1,2,5-trichloronaphthalene $C_{10}H_5Cl_3$ (PCN-15) [55720-33-7] MMHZKSMAFILONG-UHFFFAOYSA-N | $1.1\times10^{-1}$ | 5300 | Odabasi and Adali (2016) | M | 718 |
| 1,2,6-trichloronaphthalene $C_{10}H_5Cl_3$ (PCN-16) [51570-44-6] AAUJSCRITBXDJH-UHFFFAOYSA-N | $1.7\times10^{-1}$ | 6500 | Odabasi and Adali (2016) | M | 718 |
| 1,2,7-trichloronaphthalene $C_{10}H_5Cl_3$ (PCN-17) [55720-34-8] QYYUVUXSJZJCLQ-UHFFFAOYSA-N | $1.5\times10^{-1}$ | | Ebert et al. (2023) | ? | 365 |
| 1,3,5-trichloronaphthalene $C_{10}H_5Cl_3$ (PCN-19) [51570-43-5] DZHZYPCCEISGSQ-UHFFFAOYSA-N | $7.1\times10^{-2}$ | 5500 | Odabasi and Adali (2016) | M | 718 |



Table A6.2: Polychlorinated naphthalenes (PCNs) (...continued)

| Substance Formula (Trivial Name) [CAS Registry Number] InChIKey | $H_s^{cp}$ (at $T^\ominus$) $\left[\dfrac{\text{mol}}{\text{m}^3\,\text{Pa}}\right]$ | $\dfrac{\text{d}\ln H_s^{cp}}{\text{d}(1/T)}$ [K] | Reference | Type | Note |
|---|---|---|---|---|---|
| 1,4,5-trichloronaphthalene<br>$C_{10}H_5Cl_3$<br>(PCN-23)<br>[2437-55-0]<br>VQSNXLCFFHGXTI-UHFFFAOYSA-N | $2.2\times10^{-1}$ | 5400 | Odabasi and Adali (2016) | M | 718 |
| 1,4,6-trichloronaphthalene<br>$C_{10}H_5Cl_3$<br>(PCN-24)<br>[2437-54-9]<br>RLTTZFDRZKJVKJ-UHFFFAOYSA-N | $8.3\times10^{-2}$ | | Ebert et al. (2023) | ? | 365 |
| 1,6,7-trichloronaphthalene<br>$C_{10}H_5Cl_3$<br>(PCN-25)<br>[55720-39-3]<br>FUEZTEBYLIMNHN-UHFFFAOYSA-N | $1.5\times10^{-1}$ | | Ebert et al. (2023) | ? | 365 |
| 1,2,3,4-tetrachloronaphthalene<br>$C_{10}H_4Cl_4$<br>(PCN-27)<br>[20020-02-4]<br>NAQWICRLNQSPPW-UHFFFAOYSA-N | $4.1\times10^{-2}$<br>$4.1\times10^{-2}$<br>$8.3\times10^{-2}$ | | Duchowicz et al. (2020)<br>HSDB (2015)<br>Duchowicz et al. (2020) | V<br>V<br>Q | 186 |
| 1,2,3,5-tetrachloronaphthalene<br>$C_{10}H_4Cl_4$<br>(PCN-28)<br>[53555-63-8]<br>HJJKSUVYCQAMBG-UHFFFAOYSA-N | $1.3\times10^{-1}$ | | Ebert et al. (2023) | ? | 716 |
| 1,2,4,5-tetrachloronaphthalene<br>$C_{10}H_4Cl_4$<br>(PCN-32)<br>[6733-54-6]<br>BIVDISPRSYAHQQ-UHFFFAOYSA-N | $1.5\times10^{-1}$ | 5900 | Odabasi and Adali (2016) | M | 718 |
| 1,2,4,6-tetrachloronaphthalene<br>$C_{10}H_4Cl_4$<br>(PCN-33)<br>[51570-45-7]<br>GLVVZPZGCNEVEM-UHFFFAOYSA-N | $7.8\times10^{-2}$ | | Ebert et al. (2023) | ? | 365 |
| 1,2,4,7-tetrachloronaphthalene<br>$C_{10}H_4Cl_4$<br>(PCN-34)<br>[67922-21-8]<br>PWXOBMRJWBBEED-UHFFFAOYSA-N | $7.8\times10^{-2}$ | | Ebert et al. (2023) | ? | 365 |



Table A6.2: Polychlorinated naphthalenes (PCNs) (...continued)

| Substance Formula (Trivial Name) [CAS Registry Number] InChIKey | $H_s^{cp}$ (at $T^{\ominus}$) $\left[\dfrac{\mathrm{mol}}{\mathrm{m^3\,Pa}}\right]$ | $\dfrac{\mathrm{d}\ln H_s^{cp}}{\mathrm{d}(1/T)}$ [K] | Reference | Type | Note |
|---|---|---|---|---|---|
| 1,2,4,8-tetrachloronaphthalene $C_{10}H_4Cl_4$ (PCN-35) [6529-87-9] WCMSFBRREKZZFL-UHFFFAOYSA-N | $1.8\times10^{-1}$ | 6000 | Odabasi and Adali (2016) | M | 718 |
| 1,2,5,6-tetrachloronaphthalene $C_{10}H_4Cl_4$ (PCN-36) [67922-22-9] ZPUUGNBIWHBXBM-UHFFFAOYSA-N | $1.0\times10^{-1}$ | | Ebert et al. (2023) | ? | 365 |
| 1,2,5,7-tetrachloronaphthalene $C_{10}H_4Cl_4$ (PCN-37) [67922-23-0] KQZUIOXHRIVQGR-UHFFFAOYSA-N | $7.8\times10^{-2}$ | | Ebert et al. (2023) | ? | 365 |
| 1,2,5,8-tetrachloronaphthalene $C_{10}H_4Cl_4$ (PCN-38) [149864-80-2] DLTBLLHAQLBHDR-UHFFFAOYSA-N | $2.2\times10^{-1}$ | | Ebert et al. (2023) | ? | 365 |
| 1,2,6,8-tetrachloronaphthalene $C_{10}H_4Cl_4$ (PCN-40) [67922-24-1] OVAYDYKVLHHIDQ-UHFFFAOYSA-N | $2.2\times10^{-1}$ | | Ebert et al. (2023) | ? | 365 |
| 1,3,5,7-tetrachloronaphthalene $C_{10}H_4Cl_4$ (PCN-42) [53555-64-9] OTTCXKPQKOLSJN-UHFFFAOYSA-N | $5.6\times10^{-2}$ | 5600 | Odabasi and Adali (2016) | M | 718 |
| 1,3,5,8-tetrachloronaphthalene $C_{10}H_4Cl_4$ (PCN-43) [31604-28-1] VFTLNRFRXWCJJK-UHFFFAOYSA-N | $1.3\times10^{-1}$ | | Ebert et al. (2023) | ? | 717 |
| 1,3,6,8-tetrachloronaphthalene $C_{10}H_4Cl_4$ (PCN-45) [150224-15-0] XXWQPOHDPJIYIN-UHFFFAOYSA-N | $1.0\times10^{-1}$ | | Ebert et al. (2023) | ? | 365 |



Table A6.2: Polychlorinated naphthalenes (PCNs) (. . . continued)

| Substance Formula (Trivial Name) [CAS Registry Number] InChIKey | $H_s^{cp}$ (at $T^{\ominus}$) $\left[\dfrac{\mathrm{mol}}{\mathrm{m^3\,Pa}}\right]$ | $\dfrac{\mathrm{d}\ln H_s^{cp}}{\mathrm{d}(1/T)}$ [K] | Reference | Type | Note |
|---|---|---|---|---|---|
| 1,4,5,8-tetrachloronaphthalene $C_{10}H_4Cl_4$ (PCN-46) [3432-57-3] LITCKAVLJAKHOE-UHFFFAOYSA-N | $3.5 \times 10^{-1}$ | 5700 | Odabasi and Adali (2016) | M | 718 |
| 1,4,6,7-tetrachloronaphthalene $C_{10}H_4Cl_4$ (PCN-47) [55720-43-9] VJZRCIYSYVGDMU-UHFFFAOYSA-N | $9.2 \times 10^{-2}$ | 5800 | Odabasi and Adali (2016) | M | 718 |
| 1,2,3,4,6-pentachloronaphthalene $C_{10}H_3Cl_5$ (PCN-50) [67922-26-3] BAOLNVSMVTYGDA-UHFFFAOYSA-N | $7.8 \times 10^{-2}$ | 9000 | Odabasi and Adali (2016) | M | 718 |
| 1,2,3,5,7-pentachloronaphthalene $C_{10}H_3Cl_5$ (PCN-52) [53555-65-0] OVSKLQPHXHPXDR-UHFFFAOYSA-N | $6.9 \times 10^{-2}$ | | Ebert et al. (2023) | ? | 365 |
| 1,2,3,5,8-pentachloronaphthalene $C_{10}H_3Cl_5$ (PCN-53) [150224-24-1] HVYRFNJXZVEGFK-UHFFFAOYSA-N | $1.7 \times 10^{-1}$ | 9700 | Odabasi and Adali (2016) | M | 718 |
| 1,2,4,5,6-pentachloronaphthalene $C_{10}H_3Cl_5$ (PCN-57) [150224-20-7] KPOZENRUJCHWOA-UHFFFAOYSA-N | $1.4 \times 10^{-1}$ | 9500 | Odabasi and Adali (2016) | M | 718 |
| 1,2,4,5,7-pentachloronaphthalene $C_{10}H_3Cl_5$ (PCN-58) [150224-19-4] WYLDWCYZCFRVRH-UHFFFAOYSA-N | $8.9 \times 10^{-2}$ | 8700 | Odabasi and Adali (2016) | M | 718 |
| 1,2,4,5,8-pentachloronaphthalene $C_{10}H_3Cl_5$ (PCN-59) [150224-25-2] FEIKEVSWLMYFFF-UHFFFAOYSA-N | $2.6 \times 10^{-1}$ | 9100 | Odabasi and Adali (2016) | M | 718 |



Table A6.2: Polychlorinated naphthalenes (PCNs) (...continued)

| Substance<br>Formula<br>(Trivial Name)<br>[CAS Registry Number]<br>InChIKey | $H_s^{cp}$<br>(at $T^{\ominus}$)<br>$\left[\dfrac{\text{mol}}{\text{m}^3\,\text{Pa}}\right]$ | $\dfrac{\text{d}\ln H_s^{cp}}{\text{d}(1/T)}$<br><br>[K] | Reference | Type | Note |
|---|---|---|---|---|---|
| 1,2,4,6,7-pentachloronaphthalene<br>$C_{10}H_3Cl_5$<br>(PCN-60)<br>[150224-17-2]<br>GXQUDLBNLKOIQB-UHFFFAOYSA-N | $6.9\times10^{-2}$ | | Ebert et al. (2023) | ? | 365 |
| 1,2,4,6,8-pentachloronaphthalene<br>$C_{10}H_3Cl_5$<br>(PCN-61)<br>[150224-22-9]<br>HGSDQSUMXKHGTH-UHFFFAOYSA-N | $1.0\times10^{-1}$ | 8800 | Odabasi and Adali (2016) | M | 718 |
| 1,2,4,7,8-pentachloronaphthalene<br>$C_{10}H_3Cl_5$<br>(PCN-62)<br>[150224-21-8]<br>LBCOXKFWBDTJTF-UHFFFAOYSA-N | $1.7\times10^{-1}$ | 9200 | Odabasi and Adali (2016) | M | 718 |
| 1,2,3,4,5,6-hexachloronaphthalene<br>$C_{10}H_2Cl_6$<br>(PCN-63)<br>[58877-88-6]<br>CTLMCQOGOWNFHA-UHFFFAOYSA-N | $7.8\times10^{-1}$ | 12000 | Odabasi and Adali (2016) | M | 718 |
| 1,2,3,4,5,7-hexachloronaphthalene<br>$C_{10}H_2Cl_6$<br>(PCN-64)<br>[67922-27-4]<br>SWRNUKWDDYPZGV-UHFFFAOYSA-N | $3.7\times10^{-1}$ | | Ebert et al. (2023) | ? | 365 |
| 1,2,3,4,5,8-hexachloronaphthalene<br>$C_{10}H_2Cl_6$<br>(PCN-65)<br>[103426-93-3]<br>PGCDNPCENWGYMA-UHFFFAOYSA-N | $9.3\times10^{-1}$ | 10000 | Odabasi and Adali (2016) | M | 718 |
| 1,2,3,4,6,7-hexachloronaphthalene<br>$C_{10}H_2Cl_6$<br>(PCN-66)<br>[103426-96-6]<br>ZRNSVEOEIWQEMU-UHFFFAOYSA-N | $4.3\times10^{-1}$ | | Ebert et al. (2023) | ? | 716 |
| 1,2,3,5,6,7-hexachloronaphthalene<br>$C_{10}H_2Cl_6$<br>(PCN-67)<br>[103426-97-7]<br>XZLJCGGEQLNWDT-UHFFFAOYSA-N | $2.5\times10^{-1}$ | | Ebert et al. (2023) | ? | 365 |



Table A6.2: Polychlorinated naphthalenes (PCNs) (...continued)

| Substance Formula (Trivial Name) [CAS Registry Number] InChIKey | $H_s^{cp}$ (at $T^{\ominus}$) $\left[\dfrac{\text{mol}}{\text{m}^3\,\text{Pa}}\right]$ | $\dfrac{\text{d}\ln H_s^{cp}}{\text{d}(1/T)}$ [K] | Reference | Type | Note |
|---|---|---|---|---|---|
| 1,2,3,5,6,8-hexachloronaphthalene $C_{10}H_2Cl_6$ (PCN-68) [103426-95-5] FQELOCOACCYGLL-UHFFFAOYSA-N | $3.7\times10^{-1}$ | | Ebert et al. (2023) | ? | 365 |
| 1,2,3,5,7,8-hexachloronaphthalene $C_{10}H_2Cl_6$ (PCN-69) [103426-94-4] JPQLLIUTUFJWMH-UHFFFAOYSA-N | $4.3\times10^{-1}$ | 12000 | Odabasi and Adali (2016) | M | 718 |
| 1,2,4,5,6,8-hexachloronaphthalene $C_{10}H_2Cl_6$ (PCN-71) [90948-28-0] JHKLUUFTHIWTKX-UHFFFAOYSA-N | $4.5\times10^{-1}$ | | Ebert et al. (2023) | ? | 365 |
| 1,2,4,5,7,8-hexachloronaphthalene $C_{10}H_2Cl_6$ (PCN-72) [103426-92-2] SFZREMCYQNYZMZ-UHFFFAOYSA-N | $4.5\times10^{-1}$ | | Ebert et al. (2023) | ? | 365 |
| 1,2,3,4,5,6,7-heptachloronaphthalene $C_{10}HCl_7$ (PCN-73) [58863-14-2] NDZIBNJHNBUHKW-UHFFFAOYSA-N | 3.6 | 10000 | Odabasi and Adali (2016) | M | 718 |
| 1,2,3,4,5,6,8-heptachloronaphthalene $C_{10}HCl_7$ (PCN-74) [58863-15-3] QYEGXUUXWMKHHS-UHFFFAOYSA-N | 2.4 | 10000 | Odabasi and Adali (2016) | M | 718 |
| octachloronaphthalene $C_{10}Cl_8$ (PCN-75) [2234-13-1] RTNLUFLDZOAXIC-UHFFFAOYSA-N | 1.2 $1.4\times10^{-2}$ $1.4\times10^{-2}$ $7.8\times10^{-2}$ | 7800 | Odabasi and Adali (2016) Duchowicz et al. (2020) HSDB (2015) Duchowicz et al. (2020) | M V V Q | 718 186 |





### A6.3 Polychlorinated biphenyls (PCBs)

Table A6.3: Polychlorinated biphenyls (PCBs)

| Substance Formula (Trivial Name) [CAS Registry Number] InChIKey | $H_s^{cp}$ (at $T^{\ominus}$) $\left[\dfrac{\mathrm{mol}}{\mathrm{m^3\,Pa}}\right]$ | $\dfrac{\mathrm{d\ln}H_s^{cp}}{\mathrm{d}(1/T)}$ [K] | Reference | Type | Note |
|---|---|---|---|---|---|
| 2-chlorobiphenyl | $3.0\times10^{-2}$ | | Lau et al. (2006) | M | 719 |
| $C_{12}H_9Cl$ | $2.3\times10^{-2}$ | | Lau et al. (2006) | M | 720 |
| (PCB-1) | $3.0\times10^{-2}$ | 5300 | Charles and Destaillats (2005) | M | |
| [2051-60-7] | $4.9\times10^{-2}$ | 5100 | Bamford et al. (2000) | M | |
| LAXBNTIAOJWAOP-UHFFFAOYSA-N | $1.7\times10^{-2}$ | 5300 | Paasivirta and Sinkkonen (2009) | V | |
| | $1.4\times10^{-2}$ | | Mackay et al. (2006b) | V | |
| | $1.4\times10^{-2}$ | | Mackay et al. (1992a) | V | |
| | $2.7\times10^{-3}$ | | Hwang et al. (1992) | V | |
| | $1.4\times10^{-2}$ | | Shiu and Mackay (1986) | V | |
| | $3.5\times10^{-2}$ | | Burkhard et al. (1985) | V | |
| | $4.5\times10^{-2}$ | | Keshavarz et al. (2022) | Q | |
| | $3.9\times10^{-2}$ | | Duchowicz et al. (2020) | Q | 184 |
| | $2.3\times10^{-2}$ | | Hilal et al. (2008) | Q | |
| | $7.0\times10^{-2}$ | | Modarresi et al. (2007) | Q | 67 |
| | $1.7\times10^{-2}$ | | Fang Lee (2007) | Q | 721 |
| | $2.2\times10^{-2}$ | | Fang Lee (2007) | Q | 722 |
| | | 4600 | Kühne et al. (2005) | Q | |
| | $3.3\times10^{-2}$ | | Dunnivant et al. (1992) | Q | |
| | $1.3\times10^{-2}$ | | Duchowicz et al. (2020) | ? | 185, 21 |
| | | 5400 | Kühne et al. (2005) | ? | |
| 3-chlorobiphenyl | $3.2\times10^{-2}$ | 5400 | Paasivirta and Sinkkonen (2009) | V | |
| $C_{12}H_9Cl$ | $1.3\times10^{-2}$ | | Mackay et al. (2006b) | V | |
| (PCB-2) | $1.3\times10^{-2}$ | | Mackay et al. (1992a) | V | |
| [2051-61-8] | $1.3\times10^{-2}$ | | Shiu and Mackay (1986) | V | |
| NMWSKOLWZZWHPL-UHFFFAOYSA-N | $6.9\times10^{-2}$ | | Burkhard et al. (1985) | V | |
| | $4.5\times10^{-2}$ | | Keshavarz et al. (2022) | Q | |
| | $3.9\times10^{-2}$ | | Duchowicz et al. (2020) | Q | 299 |
| | $3.7\times10^{-2}$ | | Hilal et al. (2008) | Q | |
| | $1.3\times10^{-1}$ | | Modarresi et al. (2007) | Q | 67 |
| | $3.7\times10^{-2}$ | | Fang Lee (2007) | Q | 721 |
| | $3.1\times10^{-2}$ | | Fang Lee (2007) | Q | 722 |
| | $3.5\times10^{-2}$ | | Dunnivant et al. (1992) | Q | |
| | $1.6\times10^{-2}$ | | Duchowicz et al. (2020) | ? | 185, 21 |
| 4-chlorobiphenyl | $2.8\times10^{-2}$ | 5700 | Li et al. (2003) | L | 366 |
| $C_{12}H_9Cl$ | $4.2\times10^{-2}$ | 6100 | Li et al. (2003) | L | 367 |
| (PCB-3) | $3.6\times10^{-2}$ | | Lau et al. (2006) | M | 719 |
| [2051-62-9] | $2.9\times10^{-2}$ | | Lau et al. (2006) | M | 720 |
| FPWNLURCHDRMHC-UHFFFAOYSA-N | $3.5\times10^{-2}$ | 6700 | Charles and Destaillats (2005) | M | |
| | $5.6\times10^{-2}$ | 6700 | Bamford et al. (2002) | M | |
| | $1.4\times10^{-2}$ | 5100 | Paasivirta and Sinkkonen (2009) | V | |
| | $2.3\times10^{-2}$ | | Mackay et al. (2006b) | V | |
| | $2.3\times10^{-2}$ | | Mackay et al. (1992a) | V | |
| | $2.3\times10^{-2}$ | | Shiu and Mackay (1986) | V | |
| | $7.7\times10^{-2}$ | | Burkhard et al. (1985) | V | |



Table A6.3: Polychlorinated biphenyls (PCBs) (...continued)

| Substance Formula (Trivial Name) [CAS Registry Number] InChIKey | $H_s^{cp}$ (at $T^{\ominus}$) $\left[\dfrac{\mathrm{mol}}{\mathrm{m}^3\,\mathrm{Pa}}\right]$ | $\dfrac{\mathrm{d}\ln H_s^{cp}}{\mathrm{d}(1/T)}$ [K] | Reference | Type | Note |
|---|---|---|---|---|---|
| | $4.5\times10^{-2}$ | | Keshavarz et al. (2022) | Q | |
| | $3.9\times10^{-2}$ | | Duchowicz et al. (2020) | Q | 299 |
| | $3.9\times10^{-2}$ | | Hilal et al. (2008) | Q | |
| | $1.4\times10^{-1}$ | | Modarresi et al. (2007) | Q | 67 |
| | $2.1\times10^{-2}$ | | Fang Lee (2007) | Q | 721 |
| | $3.4\times10^{-2}$ | | Fang Lee (2007) | Q | 722 |
| | $3.6\times10^{-2}$ | | Dunnivant et al. (1992) | Q | |
| | $1.7\times10^{-2}$ | | Duchowicz et al. (2020) | ? | 185, 21 |
| 2,2'-dichlorobiphenyl $C_{12}H_8Cl_2$ (PCB-4) [13029-08-8] JAYCNKDKIKZTAF-UHFFFAOYSA-N | $4.6\times10^{-2}$ | 6000 | Bamford et al. (2002) | M | |
| | $4.0\times10^{-2}$ | | Fendinger and Glotfelty (1990) | M | |
| | $2.9\times10^{-2}$ | | Dunnivant et al. (1988) | M | |
| | $2.9\times10^{-2}$ | | Dunnivant and Elzerman (1988) | M | 723 |
| | $3.3\times10^{-2}$ | | Murphy et al. (1987) | M | 12 |
| | $4.5\times10^{-2}$ | | Murphy et al. (1983a) | M | 24 |
| | $7.1\times10^{-3}$ | 5500 | Paasivirta and Sinkkonen (2009) | V | |
| | $1.7\times10^{-2}$ | | Mackay et al. (2006b) | V | |
| | $1.7\times10^{-2}$ | | Mackay et al. (1992a) | V | |
| | $1.7\times10^{-2}$ | | Shiu and Mackay (1986) | V | |
| | $1.8\times10^{-2}$ | | Burkhard et al. (1985) | V | |
| | $2.6\times10^{-2}$ | | Chiou et al. (1980) | V | |
| | $4.5\times10^{-2}$ | | Murphy et al. (1983b) | X | 724, 24 |
| | $4.5\times10^{-2}$ | | Keshavarz et al. (2022) | Q | |
| | $5.5\times10^{-2}$ | | Duchowicz et al. (2020) | Q | 299 |
| | $2.5\times10^{-2}$ | | Hilal et al. (2008) | Q | |
| | $6.9\times10^{-2}$ | | Modarresi et al. (2007) | Q | 67 |
| | $2.7\times10^{-2}$ | | Fang Lee (2007) | Q | 721 |
| | $2.1\times10^{-2}$ | | Fang Lee (2007) | Q | 722 |
| | $4.8\times10^{-2}$ | | English and Carroll (2001) | Q | 230, 231 |
| | $3.0\times10^{-2}$ | | Dunnivant et al. (1992) | Q | |
| | $4.3\times10^{-2}$ | | Duchowicz et al. (2020) | ? | 185, 21 |
| 2,3-dichlorobiphenyl $C_{12}H_8Cl_2$ (PCB-5) [16605-91-7] XOMKZKJEJBZBJJ-UHFFFAOYSA-N | $4.3\times10^{-2}$ | 5800 | Bamford et al. (2002) | M | |
| | $2.1\times10^{-2}$ | 5500 | Paasivirta and Sinkkonen (2009) | V | |
| | $5.1\times10^{-2}$ | | Burkhard et al. (1985) | V | |
| | $4.5\times10^{-2}$ | | Keshavarz et al. (2022) | Q | |
| | $7.1\times10^{-2}$ | | Duchowicz et al. (2020) | Q | 184 |
| | $4.7\times10^{-2}$ | | Hilal et al. (2008) | Q | |
| | $8.2\times10^{-2}$ | | Modarresi et al. (2007) | Q | 67 |
| | $3.8\times10^{-2}$ | | Fang Lee (2007) | Q | 721 |
| | $4.1\times10^{-2}$ | | Fang Lee (2007) | Q | 722 |
| | | 5000 | Kühne et al. (2005) | Q | |
| | $4.3\times10^{-2}$ | | Yaffe et al. (2003) | Q | 248, 272 |
| | $4.1\times10^{-2}$ | | Dunnivant et al. (1992) | Q | |
| | $3.5\times10^{-2}$ | | Sabljić and Güsten (1989) | Q | |
| | $4.3\times10^{-2}$ | | Duchowicz et al. (2020) | ? | 185, 21 |
| | | 5800 | Kühne et al. (2005) | ? | |



Table A6.3: Polychlorinated biphenyls (PCBs) (...continued)

| Substance<br>Formula<br>(Trivial Name)<br>[CAS Registry Number]<br>InChIKey | $H_s^{cp}$<br>(at $T^{\ominus}$)<br>$\left[\dfrac{\text{mol}}{\text{m}^3\,\text{Pa}}\right]$ | $\dfrac{\text{d}\ln H_s^{cp}}{\text{d}(1/T)}$<br><br>[K] | Reference | Type | Note |
|---|---|---|---|---|---|
| 2,3'-dichlorobiphenyl | $4.3\times10^{-2}$ | 5700 | Bamford et al. (2002) | M | |
| $C_{12}H_8Cl_2$ | $3.9\times10^{-2}$ | | Brunner et al. (1990) | M | |
| (PCB-6) | $3.2\times10^{-2}$ | | Murphy et al. (1987) | M | 12 |
| [25569-80-6] | $4.7\times10^{-2}$ | | Murphy et al. (1983a) | M | 24 |
| ZHBBDTRJIVXKEX-UHFFFAOYSA-N | $3.9\times10^{-2}$ | 5900 | Paasivirta and Sinkkonen (2009) | V | |
| | $2.5\times10^{-2}$ | | Shiu and Mackay (1986) | V | |
| | $3.6\times10^{-2}$ | | Burkhard et al. (1985) | V | |
| | $4.5\times10^{-2}$ | | Keshavarz et al. (2022) | Q | |
| | $5.5\times10^{-2}$ | | Duchowicz et al. (2020) | Q | 184 |
| | $5.6\times10^{-2}$ | | Hilal et al. (2008) | Q | |
| | $7.2\times10^{-2}$ | | Modarresi et al. (2007) | Q | 67 |
| | $3.4\times10^{-2}$ | | Fang Lee (2007) | Q | 721 |
| | $3.3\times10^{-2}$ | | Fang Lee (2007) | Q | 722 |
| | $4.3\times10^{-2}$ | | Yaffe et al. (2003) | Q | 248, 249 |
| | $3.0\times10^{-2}$ | | Dunnivant et al. (1992) | Q | |
| | $2.5\times10^{-2}$ | | Sabljić and Güsten (1989) | Q | |
| | $3.9\times10^{-2}$ | | Duchowicz et al. (2020) | ? | 185, 21 |
| 2,4-dichlorobiphenyl | $3.7\times10^{-2}$ | 5200 | Bamford et al. (2002) | M | |
| $C_{12}H_8Cl_2$ | $2.8\times10^{-2}$ | | Dunnivant and Elzerman (1988) | M | |
| (PCB-7) | $2.7\times10^{-2}$ | | Murphy et al. (1987) | M | 12 |
| [33284-50-3] | $3.0\times10^{-2}$ | 5700 | Paasivirta and Sinkkonen (2009) | V | |
| WEJZHZJJXPXXMU-UHFFFAOYSA-N | $2.2\times10^{-2}$ | | Mackay et al. (2006b) | V | |
| | $2.2\times10^{-2}$ | | Mackay et al. (1992a) | V | |
| | $2.2\times10^{-2}$ | | Shiu and Mackay (1986) | V | |
| | $3.4\times10^{-2}$ | | Burkhard et al. (1985) | V | |
| | $4.5\times10^{-2}$ | | Keshavarz et al. (2022) | Q | |
| | $4.2\times10^{-2}$ | | Duchowicz et al. (2020) | Q | |
| | $3.9\times10^{-2}$ | | Hilal et al. (2008) | Q | |
| | $8.6\times10^{-2}$ | | Modarresi et al. (2007) | Q | 67 |
| | $2.2\times10^{-2}$ | | Fang Lee (2007) | Q | 721 |
| | $3.0\times10^{-2}$ | | Fang Lee (2007) | Q | 722 |
| | | 4700 | Kühne et al. (2005) | Q | |
| | $3.4\times10^{-2}$ | | Yaffe et al. (2003) | Q | 248, 249 |
| | $2.6\times10^{-2}$ | | Dunnivant et al. (1992) | Q | |
| | $3.5\times10^{-2}$ | | Duchowicz et al. (2020) | ? | 185, 21 |
| | | 5500 | Kühne et al. (2005) | ? | |
| 2,4'-dichlorobiphenyl | $3.8\times10^{-2}$ | 6000 | Li et al. (2003) | L | 366 |
| $C_{12}H_8Cl_2$ | $4.4\times10^{-2}$ | 6300 | Li et al. (2003) | L | 367 |
| (PCB-8) | $2.6\times10^{-2}$ | | Lau et al. (2006) | M | 719 |
| [34883-43-7] | $1.9\times10^{-2}$ | | Lau et al. (2006) | M | 720 |
| UFNIBRDIUNVOMX-UHFFFAOYSA-N | $2.3\times10^{-2}$ | 5300 | Charles and Destaillats (2005) | M | |
| | $4.0\times10^{-2}$ | 5300 | Bamford et al. (2000) | M | |
| | $3.5\times10^{-2}$ | | Murphy et al. (1987) | M | 12 |
| | $5.7\times10^{-2}$ | | Brownawell (1986) | M | 294 |
| | $4.5\times10^{-2}$ | | Murphy et al. (1983a) | M | 24 |
| | $1.0\times10^{-2}$ | | Atlas et al. (1982) | M | 679 |





Table A6.3: Polychlorinated biphenyls (PCBs) (...continued)

| Substance Formula (Trivial Name) [CAS Registry Number] InChIKey | $H_s^{cp}$ (at $T^{\ominus}$) $\left[\dfrac{\text{mol}}{\text{m}^3\,\text{Pa}}\right]$ | $\dfrac{\text{d}\ln H_s^{cp}}{\text{d}(1/T)}$ [K] | Reference | Type | Note |
|---|---|---|---|---|---|
| | $2.2\times10^{-2}$ | 5600 | Paasivirta and Sinkkonen (2009) | V | |
| | $1.1\times10^{-2}$ | | Shiu and Mackay (1986) | V | |
| | $4.0\times10^{-2}$ | | Burkhard et al. (1985) | V | |
| | $4.5\times10^{-2}$ | | Keshavarz et al. (2022) | Q | |
| | $5.5\times10^{-2}$ | | Duchowicz et al. (2020) | Q | 299 |
| | $5.7\times10^{-2}$ | | Hilal et al. (2008) | Q | |
| | $8.4\times10^{-2}$ | | Modarresi et al. (2007) | Q | 67 |
| | $1.7\times10^{-2}$ | | Fang Lee (2007) | Q | 721 |
| | $3.4\times10^{-2}$ | | Fang Lee (2007) | Q | 722 |
| | | 4700 | Kühne et al. (2005) | Q | |
| | $4.3\times10^{-2}$ | | Yaffe et al. (2003) | Q | 248, 249 |
| | $3.3\times10^{-2}$ | | Dunnivant et al. (1992) | Q | |
| | $3.2\times10^{-2}$ | | Sabljić and Güsten (1989) | Q | |
| | $4.3\times10^{-2}$ | | Duchowicz et al. (2020) | ? | 185, 21 |
| | | 5600 | Kühne et al. (2005) | ? | |
| 2,5-dichlorobiphenyl $C_{12}H_8Cl_2$ (PCB-9) [34883-39-1] KKQWHYGECTYFIA-UHFFFAOYSA-N | $2.3\times10^{-2}$ | 5700 | ten Hulscher et al. (1992) | M | |
| | $2.5\times10^{-2}$ | | Dunnivant et al. (1988) | M | |
| | $2.5\times10^{-2}$ | | Dunnivant and Elzerman (1988) | M | 723 |
| | $2.0\times10^{-2}$ | 5600 | Paasivirta and Sinkkonen (2009) | V | |
| | $5.0\times10^{-2}$ | | Mackay et al. (2006b) | V | |
| | $5.0\times10^{-2}$ | | Mackay et al. (1992a) | V | |
| | $5.0\times10^{-2}$ | | Shiu and Mackay (1986) | V | |
| | $3.0\times10^{-2}$ | | Burkhard et al. (1985) | V | |
| | $4.5\times10^{-2}$ | | Keshavarz et al. (2022) | Q | |
| | $3.8\times10^{-2}$ | | Duchowicz et al. (2020) | Q | 184 |
| | $4.1\times10^{-2}$ | | Hilal et al. (2008) | Q | |
| | $7.0\times10^{-2}$ | | Modarresi et al. (2007) | Q | 67 |
| | $3.1\times10^{-2}$ | | Fang Lee (2007) | Q | 721 |
| | $2.6\times10^{-2}$ | | Fang Lee (2007) | Q | 722 |
| | | 4700 | Kühne et al. (2005) | Q | |
| | $3.4\times10^{-2}$ | | Yaffe et al. (2003) | Q | 248, 249 |
| | $3.0\times10^{-2}$ | | Dunnivant et al. (1992) | Q | |
| | $3.5\times10^{-2}$ | | Duchowicz et al. (2020) | ? | 185, 21 |
| | | 5800 | Kühne et al. (2005) | ? | |
| 2,6-dichlorobiphenyl $C_{12}H_8Cl_2$ (PCB-10) [33146-45-1] IYZWUWBAFUBNCH-UHFFFAOYSA-N | $1.2\times10^{-2}$ | 5700 | Paasivirta and Sinkkonen (2009) | V | |
| | $2.1\times10^{-2}$ | | Burkhard et al. (1985) | V | |
| | $4.5\times10^{-2}$ | | Keshavarz et al. (2022) | Q | |
| | $4.2\times10^{-2}$ | | Duchowicz et al. (2020) | Q | 299 |
| | $3.0\times10^{-2}$ | | Hilal et al. (2008) | Q | |
| | $5.8\times10^{-2}$ | | Modarresi et al. (2007) | Q | 67 |
| | $1.9\times10^{-2}$ | | Fang Lee (2007) | Q | 721 |
| | $2.3\times10^{-2}$ | | Fang Lee (2007) | Q | 722 |
| | $4.3\times10^{-2}$ | | Yaffe et al. (2003) | Q | 248, 272 |
| | $2.3\times10^{-2}$ | | Dunnivant et al. (1992) | Q | |
| | $2.1\times10^{-2}$ | | Sabljić and Güsten (1989) | Q | |
| | $4.3\times10^{-2}$ | | Duchowicz et al. (2020) | ? | 185, 21 |





Table A6.3: Polychlorinated biphenyls (PCBs) (. . . continued)

| Substance<br>Formula<br>(Trivial Name)<br>[CAS Registry Number]<br>InChIKey | $H_s^{cp}$<br>(at $T^\ominus$)<br>$\left[\dfrac{\text{mol}}{\text{m}^3\,\text{Pa}}\right]$ | $\dfrac{\text{d}\ln H_s^{cp}}{\text{d}(1/T)}$<br><br>[K] | Reference | Type | Note |
|---|---|---|---|---|---|
| 3,3'-dichlorobiphenyl | $4.2\times10^{-2}$ | | Dunnivant et al. (1988) | M | |
| $C_{12}H_8Cl_2$ | $4.2\times10^{-2}$ | | Dunnivant and Elzerman (1988) | M | 723 |
| (PCB-11) | $3.4\times10^{-2}$ | 5700 | Paasivirta and Sinkkonen (2009) | V | |
| [2050-67-1] | $5.9\times10^{-2}$ | | Mackay et al. (2006b) | V | |
| KTXUOWUHFLBZPW-UHFFFAOYSA-N | $5.8\times10^{-2}$ | | Mackay et al. (1992a) | V | |
| | $7.4\times10^{-2}$ | | Burkhard et al. (1985) | V | |
| | $4.5\times10^{-2}$ | | Keshavarz et al. (2022) | Q | |
| | $5.5\times10^{-2}$ | | Duchowicz et al. (2020) | Q | 184 |
| | $9.0\times10^{-2}$ | | Hilal et al. (2008) | Q | |
| | $2.0\times10^{-1}$ | | Modarresi et al. (2007) | Q | 67 |
| | $7.4\times10^{-2}$ | | Fang Lee (2007) | Q | 721 |
| | $4.3\times10^{-2}$ | | Fang Lee (2007) | Q | 722 |
| | $3.4\times10^{-2}$ | | Dunnivant et al. (1992) | Q | |
| | $4.3\times10^{-2}$ | | Meylan and Howard (1991) | Q | |
| | $4.2\times10^{-2}$ | | Duchowicz et al. (2020) | ? | 185, 21 |
| 3,4-dichlorobiphenyl | $7.0\times10^{-2}$ | | Brunner et al. (1990) | M | |
| $C_{12}H_8Cl_2$ | $4.8\times10^{-2}$ | | Dunnivant et al. (1988) | M | |
| (PCB-12) | $4.8\times10^{-2}$ | | Dunnivant and Elzerman (1988) | M | 723 |
| [2974-92-7] | $2.0\times10^{-2}$ | 5300 | Paasivirta and Sinkkonen (2009) | V | |
| ZGHQUYZPMWMLBM-UHFFFAOYSA-N | $1.0\times10^{-1}$ | | Burkhard et al. (1985) | V | |
| | $4.5\times10^{-2}$ | | Keshavarz et al. (2022) | Q | |
| | $7.1\times10^{-2}$ | | Duchowicz et al. (2020) | Q | 184 |
| | $7.7\times10^{-2}$ | | Hilal et al. (2008) | Q | |
| | $1.1\times10^{-1}$ | | Modarresi et al. (2007) | Q | 67 |
| | $4.8\times10^{-2}$ | | Fang Lee (2007) | Q | 721 |
| | $4.3\times10^{-2}$ | | Fang Lee (2007) | Q | 722 |
| | $7.3\times10^{-2}$ | | Yaffe et al. (2003) | Q | 248, 249 |
| | $4.2\times10^{-2}$ | | Dunnivant et al. (1992) | Q | |
| | $7.0\times10^{-2}$ | | Duchowicz et al. (2020) | ? | 185, 21 |
| 3,4'-dichlorobiphenyl | $4.9\times10^{-2}$ | 6100 | Bamford et al. (2002) | M | |
| $C_{12}H_8Cl_2$ | $8.5\times10^{-2}$ | 5900 | Paasivirta and Sinkkonen (2009) | V | |
| (PCB-13) | $8.1\times10^{-2}$ | | Burkhard et al. (1985) | V | |
| [2974-90-5] | $9.5\times10^{-2}$ | | Hilal et al. (2008) | Q | |
| CJDNEKOMKXLSBN-UHFFFAOYSA-N | $3.7\times10^{-2}$ | | Fang Lee (2007) | Q | 721 |
| | $4.4\times10^{-2}$ | | Fang Lee (2007) | Q | 722 |
| | $3.9\times10^{-2}$ | | Dunnivant et al. (1992) | Q | |
| | $3.1\times10^{-2}$ | | Sabljić and Güsten (1989) | Q | |
| 3,5-dichlorobiphenyl | $2.7\times10^{-2}$ | 5500 | Paasivirta and Sinkkonen (2009) | V | |
| $C_{12}H_8Cl_2$ | $6.0\times10^{-2}$ | | Burkhard et al. (1985) | V | |
| (PCB-14) | $5.0\times10^{-2}$ | | Hilal et al. (2008) | Q | |
| [34883-41-5] | $9.4\times10^{-2}$ | | Modarresi et al. (2007) | Q | 67 |
| QHZSDTDMQZPUKC-UHFFFAOYSA-N | $6.7\times10^{-2}$ | | Fang Lee (2007) | Q | 721 |
| | $3.2\times10^{-2}$ | | Fang Lee (2007) | Q | 722 |
| | $2.3\times10^{-2}$ | | Dunnivant et al. (1992) | Q | |
| | $2.0\times10^{-2}$ | | Sabljić and Güsten (1989) | Q | |





Table A6.3: Polychlorinated biphenyls (PCBs) (…continued)

| Substance Formula (Trivial Name) [CAS Registry Number] InChIKey | $H_s^{cp}$ (at $T^\ominus$) $\left[\dfrac{\mathrm{mol}}{\mathrm{m^3\,Pa}}\right]$ | $\dfrac{\mathrm{d}\ln H_s^{cp}}{\mathrm{d}(1/T)}$ [K] | Reference | Type | Note |
|---|---|---|---|---|---|
| 4,4'-dichlorobiphenyl | $7.0\times10^{-2}$ | 6000 | Li et al. (2003) | L | 366 |
| $C_{12}H_8Cl_2$ | $7.5\times10^{-2}$ | 6700 | Li et al. (2003) | L | 367 |
| (PCB-15) | $5.0\times10^{-2}$ | | Lau et al. (2006) | M | 719 |
| [2050-68-2] | $3.3\times10^{-2}$ | | Lau et al. (2006) | M | 720 |
| YTBRNEUEFCNVHC-UHFFFAOYSA-N | $3.5\times10^{-2}$ | 5300 | Charles and Destaillats (2005) | M | 33 |
| | $1.0\times10^{-1}$ | | Fendinger and Glotfelty (1990) | M | |
| | $5.0\times10^{-2}$ | | Dunnivant et al. (1988) | M | |
| | $5.0\times10^{-2}$ | | Dunnivant and Elzerman (1988) | M | 723 |
| | $3.3\times10^{-3}$ | 4900 | Paasivirta and Sinkkonen (2009) | V | |
| | $5.6\times10^{-2}$ | | Mackay et al. (2006b) | V | |
| | $5.9\times10^{-2}$ | | Mackay et al. (1992a) | V | |
| | $5.9\times10^{-2}$ | | Shiu and Mackay (1986) | V | |
| | $9.1\times10^{-2}$ | | Burkhard et al. (1985) | V | |
| | $1.0\times10^{-1}$ | | Chiou et al. (1980) | V | |
| | $3.3\times10^{-2}$ | | Murphy et al. (1983b) | X | 724, 24 |
| | $6.8\times10^{-2}$ | | Dunnivant et al. (1988) | C | |
| | $4.5\times10^{-2}$ | | Keshavarz et al. (2022) | Q | |
| | $5.5\times10^{-2}$ | | Duchowicz et al. (2020) | Q | 184 |
| | $9.7\times10^{-2}$ | | Hilal et al. (2008) | Q | |
| | $1.8\times10^{-1}$ | | Modarresi et al. (2007) | Q | 67 |
| | $2.1\times10^{-2}$ | | Fang Lee (2007) | Q | 721 |
| | $4.8\times10^{-2}$ | | Fang Lee (2007) | Q | 722 |
| | $4.4\times10^{-2}$ | | Dunnivant et al. (1992) | Q | |
| | $4.3\times10^{-2}$ | | Meylan and Howard (1991) | Q | |
| | $5.0\times10^{-2}$ | | Duchowicz et al. (2020) | ? | 185, 21 |
| 2,2',3-trichlorobiphenyl | $4.2\times10^{-2}$ | 5700 | Bamford et al. (2002) | M | |
| $C_{12}H_7Cl_3$ | $4.1\times10^{-2}$ | | Murphy et al. (1987) | M | 12 |
| (PCB-16) | $1.2\times10^{-2}$ | | Atlas et al. (1982) | M | 679 |
| [38444-78-9] | $1.5\times10^{-2}$ | 5800 | Paasivirta and Sinkkonen (2009) | V | |
| XVIZMMSINIOIQP-UHFFFAOYSA-N | $1.3\times10^{-2}$ | | Shiu and Mackay (1986) | V | |
| | $2.8\times10^{-2}$ | | Burkhard et al. (1985) | V | |
| | $4.5\times10^{-2}$ | | Keshavarz et al. (2022) | Q | |
| | $1.0\times10^{-1}$ | | Duchowicz et al. (2020) | Q | 299 |
| | $5.6\times10^{-2}$ | | Hilal et al. (2008) | Q | |
| | $6.9\times10^{-2}$ | | Modarresi et al. (2007) | Q | 67 |
| | $6.1\times10^{-2}$ | | Fang Lee (2007) | Q | 721 |
| | $4.4\times10^{-2}$ | | Fang Lee (2007) | Q | 722 |
| | | 4500 | Kühne et al. (2005) | Q | |
| | $3.9\times10^{-2}$ | | Dunnivant et al. (1992) | Q | |
| | $3.6\times10^{-2}$ | | Sabljić and Güsten (1989) | Q | |
| | $4.9\times10^{-2}$ | | Duchowicz et al. (2020) | ? | 185, 21 |
| | | 4700 | Kühne et al. (2005) | ? | |





Table A6.3: Polychlorinated biphenyls (PCBs) (...continued)

| Substance Formula (Trivial Name) [CAS Registry Number] InChIKey | $H_s^{cp}$ (at $T^\ominus$) $\left[\dfrac{\mathrm{mol}}{\mathrm{m^3\,Pa}}\right]$ | $\dfrac{\mathrm{d}\ln H_s^{cp}}{\mathrm{d}(1/T)}$ [K] | Reference | Type | Note |
|---|---|---|---|---|---|
| 2,2',4-trichlorobiphenyl | $3.3\times10^{-2}$ | 4700 | Bamford et al. (2002) | M | |
| $C_{12}H_7Cl_3$ | $3.0\times10^{-2}$ | | Murphy et al. (1987) | M | 12 |
| (PCB-17) | $4.0\times10^{-2}$ | 6200 | Paasivirta and Sinkkonen (2009) | V | |
| [37680-66-3] | $1.9\times10^{-2}$ | | Burkhard et al. (1985) | V | |
| YKKYCYQDUUXNLN-UHFFFAOYSA-N | $3.7\times10^{-2}$ | | Hilal et al. (2008) | Q | |
| | $3.5\times10^{-2}$ | | Fang Lee (2007) | Q | 721 |
| | $3.2\times10^{-2}$ | | Fang Lee (2007) | Q | 722 |
| | $2.6\times10^{-2}$ | | Dunnivant et al. (1992) | Q | |
| | $2.5\times10^{-2}$ | | Sabljić and Güsten (1989) | Q | |
| 2,2',5-trichlorobiphenyl | $2.9\times10^{-1}$ | | Bhangare et al. (2019) | M | 725 |
| $C_{12}H_7Cl_3$ | $9.0\times10^{-2}$ | | Bhangare et al. (2019) | M | 726 |
| (PCB-18) | $3.9\times10^{-2}$ | 4200 | Bamford et al. (2000) | M | |
| [37680-65-2] | $3.9\times10^{-2}$ | | Brunner et al. (1990) | M | |
| DCMURXAZTZQAFB-UHFFFAOYSA-N | $2.6\times10^{-2}$ | | Dunnivant and Elzerman (1988) | M | |
| | $3.3\times10^{-2}$ | | Murphy et al. (1987) | M | 12 |
| | $4.9\times10^{-2}$ | | Oliver (1985) | M | |
| | $4.9\times10^{-2}$ | | Murphy et al. (1983a) | M | 24 |
| | $9.9\times10^{-3}$ | | Atlas et al. (1982) | M | 679 |
| | $9.8\times10^{-3}$ | 5800 | Paasivirta and Sinkkonen (2009) | V | |
| | $1.1\times10^{-2}$ | | Mackay et al. (2006b) | V | |
| | $1.1\times10^{-2}$ | | Mackay et al. (1992a) | V | |
| | $1.1\times10^{-2}$ | | Shiu and Mackay (1986) | V | |
| | $1.7\times10^{-2}$ | | Burkhard et al. (1985) | V | |
| | $4.5\times10^{-2}$ | | Keshavarz et al. (2022) | Q | |
| | $5.5\times10^{-2}$ | | Duchowicz et al. (2020) | Q | 299 |
| | $4.6\times10^{-2}$ | | Hilal et al. (2008) | Q | |
| | $6.4\times10^{-2}$ | | Modarresi et al. (2007) | Q | 67 |
| | $4.9\times10^{-2}$ | | Fang Lee (2007) | Q | 721 |
| | $3.0\times10^{-2}$ | | Fang Lee (2007) | Q | 722 |
| | | 4200 | Kühne et al. (2005) | Q | |
| | $5.8\times10^{-2}$ | | Yaffe et al. (2003) | Q | 248, 272 |
| | $6.0\times10^{-2}$ | | English and Carroll (2001) | Q | 230, 274 |
| | $3.1\times10^{-2}$ | | Dunnivant et al. (1992) | Q | |
| | $3.9\times10^{-2}$ | | Duchowicz et al. (2020) | ? | 185, 21 |
| | | 4500 | Kühne et al. (2005) | ? | |
| 2,2',6-trichlorobiphenyl | $3.3\times10^{-2}$ | 4700 | Bamford et al. (2002) | M | |
| $C_{12}H_7Cl_3$ | $4.3\times10^{-2}$ | | Brunner et al. (1990) | M | |
| (PCB-19) | $3.3\times10^{-2}$ | | Murphy et al. (1987) | M | 12 |
| [38444-73-4] | $4.7\times10^{-2}$ | | Murphy et al. (1983a) | M | 24 |
| MVXIJRBBCDLNLX-UHFFFAOYSA-N | $2.5\times10^{-3}$ | 5400 | Paasivirta and Sinkkonen (2009) | V | |
| | $8.0\times10^{-3}$ | | Burkhard et al. (1985) | V | |
| | $4.5\times10^{-2}$ | | Keshavarz et al. (2022) | Q | |
| | $6.0\times10^{-2}$ | | Duchowicz et al. (2020) | Q | |
| | $5.4\times10^{-2}$ | | Hilal et al. (2008) | Q | |
| | $6.0\times10^{-2}$ | | Modarresi et al. (2007) | Q | 67 |
| | $3.0\times10^{-2}$ | | Fang Lee (2007) | Q | 721 |





Table A6.3: Polychlorinated biphenyls (PCBs) (. . . continued)

| Substance Formula (Trivial Name) [CAS Registry Number] InChIKey | $H_s^{cp}$ (at $T^\ominus$) $\left[\dfrac{\text{mol}}{\text{m}^3\,\text{Pa}}\right]$ | $\dfrac{\text{d}\ln H_s^{cp}}{\text{d}(1/T)}$ [K] | Reference | Type | Note |
|---|---|---|---|---|---|
| | $2.4\times10^{-2}$ | | Fang Lee (2007) | Q | 722 |
| | | 3600 | Kühne et al. (2005) | Q | |
| | $2.2\times10^{-2}$ | | Dunnivant et al. (1992) | Q | |
| | $2.2\times10^{-2}$ | | Sabljić and Güsten (1989) | Q | |
| | $4.3\times10^{-2}$ | | Duchowicz et al. (2020) | ? | 185, 21 |
| | | 3100 | Kühne et al. (2005) | ? | |
| 2,3,3'-trichlorobiphenyl $C_{12}H_7Cl_3$ (PCB-20) [38444-84-7] SXHLTVKPNQVZGL-UHFFFAOYSA-N | $1.2\times10^{-2}$ | | Atlas et al. (1982) | M | 679 |
| | $2.0\times10^{-2}$ | 5800 | Paasivirta and Sinkkonen (2009) | V | |
| | $1.2\times10^{-2}$ | | Shiu and Mackay (1986) | V | |
| | $5.8\times10^{-2}$ | | Burkhard et al. (1985) | V | |
| | $4.5\times10^{-2}$ | | Keshavarz et al. (2022) | Q | |
| | $1.0\times10^{-1}$ | | Duchowicz et al. (2020) | Q | 299 |
| | $1.2\times10^{-1}$ | | Hilal et al. (2008) | Q | |
| | $8.4\times10^{-2}$ | | Modarresi et al. (2007) | Q | 67 |
| | $7.6\times10^{-2}$ | | Fang Lee (2007) | Q | 721 |
| | $6.5\times10^{-2}$ | | Fang Lee (2007) | Q | 722 |
| | $4.5\times10^{-2}$ | | Dunnivant et al. (1992) | Q | |
| | $3.3\times10^{-2}$ | | Sabljić and Güsten (1989) | Q | |
| | $6.2\times10^{-2}$ | | Duchowicz et al. (2020) | ? | 185, 21 |
| 2,3,4-trichlorobiphenyl $C_{12}H_7Cl_3$ (PCB-21) [55702-46-0] IUYHQGMDSZOPDZ-UHFFFAOYSA-N | $4.3\times10^{-3}$ | 5200 | Paasivirta and Sinkkonen (2009) | V | |
| | $6.8\times10^{-2}$ | | Burkhard et al. (1985) | V | |
| | $7.9\times10^{-2}$ | | Hilal et al. (2008) | Q | |
| | $5.0\times10^{-2}$ | | Fang Lee (2007) | Q | 721 |
| | $5.9\times10^{-2}$ | | Fang Lee (2007) | Q | 722 |
| | $4.3\times10^{-2}$ | | Dunnivant et al. (1992) | Q | |
| | $4.7\times10^{-2}$ | | Sabljić and Güsten (1989) | Q | |
| 2,3,4'-trichlorobiphenyl $C_{12}H_7Cl_3$ (PCB-22) [38444-85-8] ZMHWQAHZKUPENF-UHFFFAOYSA-N | $3.4\times10^{-2}$ | 4800 | Bamford et al. (2002) | M | |
| | $5.0\times10^{-2}$ | | Murphy et al. (1987) | M | 12 |
| | $5.5\times10^{-2}$ | | Murphy et al. (1983a) | M | 24 |
| | $1.3\times10^{-2}$ | 5600 | Paasivirta and Sinkkonen (2009) | V | |
| | $6.5\times10^{-2}$ | | Burkhard et al. (1985) | V | |
| | $4.5\times10^{-2}$ | | Keshavarz et al. (2022) | Q | |
| | $1.0\times10^{-1}$ | | Duchowicz et al. (2020) | Q | 184 |
| | $1.2\times10^{-1}$ | | Hilal et al. (2008) | Q | |
| | $1.0\times10^{-1}$ | | Modarresi et al. (2007) | Q | 67 |
| | $3.8\times10^{-2}$ | | Fang Lee (2007) | Q | 721 |
| | $6.4\times10^{-2}$ | | Fang Lee (2007) | Q | 722 |
| | $5.2\times10^{-2}$ | | Dunnivant et al. (1992) | Q | |
| | $4.4\times10^{-2}$ | | Sabljić and Güsten (1989) | Q | |
| | $7.0\times10^{-2}$ | | Duchowicz et al. (2020) | ? | 185, 21 |





Table A6.3: Polychlorinated biphenyls (PCBs) (...continued)

| Substance<br>Formula<br>(Trivial Name)<br>[CAS Registry Number]<br>InChIKey | $H_s^{cp}$<br>(at $T^\ominus$)<br>$\left[\dfrac{\mathrm{mol}}{\mathrm{m^3\,Pa}}\right]$ | $\dfrac{\mathrm{d}\ln H_s^{cp}}{\mathrm{d}(1/T)}$<br><br>[K] | Reference | Type | Note |
|---|---|---|---|---|---|
| 2,3,5-trichlorobiphenyl | $1.5\times10^{-2}$ | 5700 | Paasivirta and Sinkkonen (2009) | V | |
| $C_{12}H_7Cl_3$ | $3.9\times10^{-2}$ | | Burkhard et al. (1985) | V | |
| (PCB-23) | $6.9\times10^{-2}$ | | Fang Lee (2007) | Q | 721 |
| [55720-44-0] | $4.7\times10^{-2}$ | | Fang Lee (2007) | Q | 722 |
| GBUCDGDROYMOAN-UHFFFAOYSA-N | $3.1\times10^{-2}$ | | Dunnivant et al. (1992) | Q | |
| | $2.8\times10^{-2}$ | | Sabljić and Güsten (1989) | Q | |
| 2,3,6-trichlorobiphenyl | $3.3\times10^{-2}$ | 4700 | Bamford et al. (2002) | M | |
| $C_{12}H_7Cl_3$ | $4.5\times10^{-2}$ | | Brunner et al. (1990) | M | |
| (PCB-24) | $3.1\times10^{-2}$ | | Murphy et al. (1987) | M | 12 |
| [55702-45-9] | $7.7\times10^{-3}$ | 5600 | Paasivirta and Sinkkonen (2009) | V | |
| LVROLHVSYNLFBE-UHFFFAOYSA-N | $2.3\times10^{-2}$ | | Burkhard et al. (1985) | V | |
| | $4.5\times10^{-2}$ | | Keshavarz et al. (2022) | Q | |
| | $5.3\times10^{-2}$ | | Duchowicz et al. (2020) | Q | 299 |
| | $4.7\times10^{-2}$ | | Hilal et al. (2008) | Q | |
| | $8.5\times10^{-2}$ | | Modarresi et al. (2007) | Q | 67 |
| | $4.2\times10^{-2}$ | | Fang Lee (2007) | Q | 721 |
| | $4.4\times10^{-2}$ | | Fang Lee (2007) | Q | 722 |
| | | 4500 | Kühne et al. (2005) | Q | |
| | $3.2\times10^{-2}$ | | Dunnivant et al. (1992) | Q | |
| | $2.9\times10^{-2}$ | | Sabljić and Güsten (1989) | Q | |
| | $4.5\times10^{-2}$ | | Duchowicz et al. (2020) | ? | 185, 21 |
| | | 2800 | Kühne et al. (2005) | ? | |
| 2,3',4-trichlorobiphenyl | $3.3\times10^{-2}$ | 4700 | Bamford et al. (2002) | M | |
| $C_{12}H_7Cl_3$ | $2.4\times10^{-2}$ | | Murphy et al. (1987) | M | 12 |
| (PCB-25) | $6.2\times10^{-2}$ | | Murphy et al. (1983a) | M | 24 |
| [55712-37-3] | $2.8\times10^{-2}$ | 5900 | Paasivirta and Sinkkonen (2009) | V | |
| XBBZAULFUPBZSP-UHFFFAOYSA-N | $3.9\times10^{-2}$ | | Burkhard et al. (1985) | V | |
| | $4.4\times10^{-2}$ | | Fang Lee (2007) | Q | 721 |
| | $4.5\times10^{-2}$ | | Fang Lee (2007) | Q | 722 |
| | | 4800 | Kühne et al. (2005) | Q | |
| | $3.1\times10^{-2}$ | | Dunnivant et al. (1992) | Q | |
| | $2.3\times10^{-2}$ | | Sabljić and Güsten (1989) | Q | |
| | | 5700 | Kühne et al. (2005) | ? | |
| 2,3',5-trichlorobiphenyl | $3.5\times10^{-2}$ | 4900 | Bamford et al. (2002) | M | |
| $C_{12}H_7Cl_3$ | $3.0\times10^{-2}$ | | Dunnivant et al. (1988) | M | |
| (PCB-26) | $3.0\times10^{-2}$ | | Dunnivant and Elzerman (1988) | M | 723 |
| [38444-81-4] | $2.9\times10^{-2}$ | | Murphy et al. (1987) | M | 12 |
| ONNCPBRWFSKDMQ-UHFFFAOYSA-N | $2.2\times10^{-2}$ | 5900 | Paasivirta and Sinkkonen (2009) | V | |
| | $3.5\times10^{-2}$ | | Burkhard et al. (1985) | V | |
| | $4.5\times10^{-2}$ | | Keshavarz et al. (2022) | Q | |
| | $5.5\times10^{-2}$ | | Duchowicz et al. (2020) | Q | 299 |
| | $9.9\times10^{-2}$ | | Hilal et al. (2008) | Q | |
| | $7.3\times10^{-2}$ | | Modarresi et al. (2007) | Q | 67 |
| | $6.1\times10^{-2}$ | | Fang Lee (2007) | Q | 721 |
| | $4.3\times10^{-2}$ | | Fang Lee (2007) | Q | 722 |
| | $3.3\times10^{-2}$ | | Dunnivant et al. (1992) | Q | |



Table A6.3: Polychlorinated biphenyls (PCBs) (. . . continued)

| Substance Formula (Trivial Name) [CAS Registry Number] InChIKey | $H_s^{cp}$ (at $T^{\ominus}$) $\left[\dfrac{\text{mol}}{\text{m}^3\,\text{Pa}}\right]$ | $\dfrac{\text{d}\ln H_s^{cp}}{\text{d}(1/T)}$ [K] | Reference | Type | Note |
|---|---|---|---|---|---|
| | $5.9\times10^{-2}$ | | Meylan and Howard (1991) | Q | |
| | $4.9\times10^{-2}$ | | Duchowicz et al. (2020) | ? | 185, 21 |
| 2,3',6-trichlorobiphenyl | $3.5\times10^{-2}$ | | Murphy et al. (1987) | M | 12 |
| $C_{12}H_7Cl_3$ | $3.1\times10^{-2}$ | 6100 | Paasivirta and Sinkkonen (2009) | V | |
| (PCB-27) | $2.4\times10^{-2}$ | | Burkhard et al. (1985) | V | |
| [38444-76-7] | $3.7\times10^{-2}$ | | Fang Lee (2007) | Q | 721 |
| VQOFJPFYTCHPTR-UHFFFAOYSA-N | $4.2\times10^{-2}$ | | Fang Lee (2007) | Q | 722 |
| | $2.4\times10^{-2}$ | | Dunnivant et al. (1992) | Q | |
| | $2.0\times10^{-2}$ | | Sabljić and Güsten (1989) | Q | |
| 2,4,4'-trichlorobiphenyl | $3.0\times10^{-2}$ | 6300 | Li et al. (2003) | L | 366 |
| $C_{12}H_7Cl_3$ | $3.3\times10^{-2}$ | 6600 | Li et al. (2003) | L | 367 |
| (PCB-28) | $5.9\times10^{-3}$ | | Bhangare et al. (2019) | M | 725 |
| [7012-37-5] | $5.0\times10^{-2}$ | | Bhangare et al. (2019) | M | 726 |
| BZTYNSQSZHARAZ-UHFFFAOYSA-N | $2.3\times10^{-2}$ | | Lau et al. (2006) | M | 719 |
| | $1.4\times10^{-2}$ | | Lau et al. (2006) | M | 720 |
| | $1.8\times10^{-2}$ | 2300 | Charles and Destaillats (2005) | M | 33 |
| | $2.6\times10^{-2}$ | 3900 | Bamford et al. (2000) | M | |
| | $3.6\times10^{-2}$ | 6100 | ten Hulscher et al. (1992) | M | |
| | $4.9\times10^{-2}$ | | Brunner et al. (1990) | M | |
| | $3.1\times10^{-2}$ | | Dunnivant and Elzerman (1988) | M | |
| | $3.7\times10^{-2}$ | | Murphy et al. (1987) | M | 12 |
| | $6.9\times10^{-2}$ | | Brownawell (1986) | M | 294 |
| | $2.7\times10^{-2}$ | 5900 | Paasivirta and Sinkkonen (2009) | V | |
| | $4.4\times10^{-2}$ | | Burkhard et al. (1985) | V | |
| | $2.7\times10^{-2}$ | 7100 | Paasivirta et al. (1999) | T | |
| | $4.5\times10^{-2}$ | | Keshavarz et al. (2022) | Q | |
| | $6.0\times10^{-2}$ | | Duchowicz et al. (2020) | Q | |
| | $1.0\times10^{-1}$ | | Hilal et al. (2008) | Q | |
| | $9.9\times10^{-2}$ | | Modarresi et al. (2007) | Q | 67 |
| | $2.2\times10^{-2}$ | | Fang Lee (2007) | Q | 721 |
| | $4.7\times10^{-2}$ | | Fang Lee (2007) | Q | 722 |
| | | 4800 | Kühne et al. (2005) | Q | |
| | $5.2\times10^{-2}$ | | Yaffe et al. (2003) | Q | 248, 249 |
| | $3.5\times10^{-2}$ | | Dunnivant et al. (1992) | Q | |
| | $4.9\times10^{-2}$ | | Duchowicz et al. (2020) | ? | 185, 21 |
| | | 4800 | Kühne et al. (2005) | ? | |
| 2,4,5-trichlorobiphenyl | $3.1\times10^{-2}$ | 6500 | Schwardt et al. (2021) | L | 1 |
| $C_{12}H_7Cl_3$ | $3.1\times10^{-2}$ | 6300 | Li et al. (2003) | L | 366 |
| (PCB-29) | $3.3\times10^{-2}$ | 6700 | Li et al. (2003) | L | 367 |
| [15862-07-4] | $2.6\times10^{-2}$ | 4200 | Bamford et al. (2000) | M | |
| VGVIKVCCUATMNG-UHFFFAOYSA-N | $4.9\times10^{-2}$ | | Brunner et al. (1990) | M | |
| | $7.7\times10^{-3}$ | 5500 | Paasivirta and Sinkkonen (2009) | V | |
| | $4.1\times10^{-3}$ | | Mackay et al. (2006b) | V | |
| | $4.1\times10^{-2}$ | | Mackay et al. (1992a) | V | |
| | $4.2\times10^{-2}$ | | Shiu and Mackay (1986) | V | |
| | $3.9\times10^{-2}$ | | Burkhard et al. (1985) | V | |



Table A6.3: Polychlorinated biphenyls (PCBs) (. . . continued)

| Substance<br>Formula<br>(Trivial Name)<br>[CAS Registry Number]<br>InChIKey | $H_s^{cp}$ (at $T^{\ominus}$) $\left[\dfrac{\text{mol}}{\text{m}^3\,\text{Pa}}\right]$ | $\dfrac{\text{d}\ln H_s^{cp}}{\text{d}(1/T)}$ [K] | Reference | Type | Note |
|---|---|---|---|---|---|
| | $4.5\times10^{-2}$ | | Keshavarz et al. (2022) | Q | |
| | $5.3\times10^{-2}$ | | Duchowicz et al. (2020) | Q | |
| | $4.5\times10^{-2}$ | | Bhangare et al. (2019) | Q | |
| | $7.9\times10^{-2}$ | | Hilal et al. (2008) | Q | |
| | $1.3\times10^{-1}$ | | Modarresi et al. (2007) | Q | 67 |
| | $4.0\times10^{-2}$ | | Fang Lee (2007) | Q | 721 |
| | $4.2\times10^{-2}$ | | Fang Lee (2007) | Q | 722 |
| | | 5100 | Kühne et al. (2005) | Q | |
| | $3.3\times10^{-2}$ | | Dunnivant et al. (1992) | Q | |
| | $3.7\times10^{-2}$ | | Sabljić and Güsten (1989) | Q | |
| | $4.9\times10^{-2}$ | | Duchowicz et al. (2020) | ? | 185, 21 |
| | | 4500 | Kühne et al. (2005) | ? | |
| 2,4,6-trichlorobiphenyl<br>$C_{12}H_7Cl_3$<br>(PCB-30)<br>[35693-92-6]<br>MTLMVEWEYZFYTH-UHFFFAOYSA-N | $1.5\times10^{-2}$ | | Dunnivant et al. (1988) | M | |
| | $1.5\times10^{-2}$ | | Dunnivant and Elzerman (1988) | M | 723 |
| | $1.0\times10^{-2}$ | | Duchowicz et al. (2020) | V | 186 |
| | $8.5\times10^{-3}$ | 5600 | Paasivirta and Sinkkonen (2009) | V | |
| | $2.0\times10^{-2}$ | | Mackay et al. (2006b) | V | |
| | $2.0\times10^{-2}$ | | Mackay et al. (1992a) | V | |
| | $2.0\times10^{-2}$ | | Shiu and Mackay (1986) | V | |
| | $1.6\times10^{-2}$ | | Burkhard et al. (1985) | V | |
| | $3.5\times10^{-2}$ | | Duchowicz et al. (2020) | Q | |
| | $2.9\times10^{-2}$ | | Hilal et al. (2008) | Q | |
| | $2.4\times10^{-2}$ | | Fang Lee (2007) | Q | 721 |
| | $2.5\times10^{-2}$ | | Fang Lee (2007) | Q | 722 |
| | $1.7\times10^{-2}$ | | Dunnivant et al. (1992) | Q | |
| 2,4',5-trichlorobiphenyl<br>$C_{12}H_7Cl_3$<br>(PCB-31)<br>[16606-02-3]<br>VAHKBZSAUKPEOV-UHFFFAOYSA-N | $2.7\times10^{-2}$ | 6100 | Li et al. (2003) | L | 366 |
| | $2.9\times10^{-2}$ | 6600 | Li et al. (2003) | L | 367 |
| | $3.4\times10^{-2}$ | 4900 | Bamford et al. (2002) | M | |
| | $5.2\times10^{-2}$ | | Brunner et al. (1990) | M | |
| | $3.7\times10^{-2}$ | | Murphy et al. (1987) | M | 12 |
| | $4.9\times10^{-2}$ | | Murphy et al. (1983a) | M | 24 |
| | $1.1\times10^{-2}$ | | Atlas et al. (1982) | M | 679 |
| | $1.3\times10^{-2}$ | 5700 | Paasivirta and Sinkkonen (2009) | V | |
| | $1.8\times10^{-2}$ | | Shiu and Mackay (1986) | V | |
| | $3.9\times10^{-2}$ | | Burkhard et al. (1985) | V | |
| | $4.5\times10^{-2}$ | | Keshavarz et al. (2022) | Q | |
| | $5.5\times10^{-2}$ | | Duchowicz et al. (2020) | Q | |
| | $1.0\times10^{-1}$ | | Hilal et al. (2008) | Q | |
| | $9.1\times10^{-2}$ | | Modarresi et al. (2007) | Q | 67 |
| | $3.0\times10^{-2}$ | | Fang Lee (2007) | Q | 721 |
| | $4.1\times10^{-2}$ | | Fang Lee (2007) | Q | 722 |
| | $3.6\times10^{-2}$ | | Dunnivant et al. (1992) | Q | |
| | $3.5\times10^{-2}$ | | Sabljić and Güsten (1989) | Q | |
| | $5.2\times10^{-2}$ | | Duchowicz et al. (2020) | ? | 185, 21 |





Table A6.3: Polychlorinated biphenyls (PCBs) (...continued)

| Substance Formula (Trivial Name) [CAS Registry Number] InChIKey | $H_s^{cp}$ (at $T^{\ominus}$) $\left[ \dfrac{\mathrm{mol}}{\mathrm{m}^3 \, \mathrm{Pa}} \right]$ | $\dfrac{\mathrm{d} \ln H_s^{cp}}{\mathrm{d}(1/T)}$ [K] | Reference | Type | Note |
|---|---|---|---|---|---|
| 2,4',6-trichlorobiphenyl | $1.2 \times 10^{-2}$ | 5700 | Paasivirta and Sinkkonen (2009) | V | |
| $C_{12}H_7Cl_3$ | $2.7 \times 10^{-2}$ | | Burkhard et al. (1985) | V | |
| (PCB-32) | $4.5 \times 10^{-2}$ | | Keshavarz et al. (2022) | Q | |
| [38444-77-8] | $6.0 \times 10^{-2}$ | | Duchowicz et al. (2020) | Q | 299 |
| IHIDFKLAWYPTKB-UHFFFAOYSA-N | $7.5 \times 10^{-2}$ | | Hilal et al. (2008) | Q | |
| | $6.9 \times 10^{-2}$ | | Modarresi et al. (2007) | Q | 67 |
| | $1.8 \times 10^{-2}$ | | Fang Lee (2007) | Q | 721 |
| | $4.1 \times 10^{-2}$ | | Fang Lee (2007) | Q | 722 |
| | $2.5 \times 10^{-2}$ | | Dunnivant et al. (1992) | Q | |
| | $2.4 \times 10^{-2}$ | | Sabljić and Güsten (1989) | Q | |
| | $4.9 \times 10^{-2}$ | | Duchowicz et al. (2020) | ? | 185, 21 |
| 2,3',4'-trichlorobiphenyl | $3.6 \times 10^{-2}$ | 5100 | Bamford et al. (2002) | M | |
| $C_{12}H_7Cl_3$ | $4.4 \times 10^{-2}$ | | Murphy et al. (1987) | M | 12 |
| (PCB-33) | $6.6 \times 10^{-2}$ | | Murphy et al. (1983a) | M | 24 |
| [38444-86-9] | $2.5 \times 10^{-2}$ | | Westcott et al. (1981) | M | |
| RIMXLXBUOQMDHV-UHFFFAOYSA-N | $1.3 \times 10^{-2}$ | 5600 | Paasivirta and Sinkkonen (2009) | V | |
| | | | Mackay et al. (2006b) | V | 683 |
| | $2.3 \times 10^{-2}$ | | Mackay et al. (1992a) | V | |
| | $2.3 \times 10^{-2}$ | | Shiu and Mackay (1986) | V | |
| | $5.9 \times 10^{-2}$ | | Burkhard et al. (1985) | V | |
| | $4.5 \times 10^{-2}$ | | Keshavarz et al. (2022) | Q | |
| | $1.0 \times 10^{-1}$ | | Duchowicz et al. (2020) | Q | 299 |
| | $1.1 \times 10^{-1}$ | | Hilal et al. (2008) | Q | |
| | $7.0 \times 10^{-2}$ | | Modarresi et al. (2007) | Q | 67 |
| | $7.7 \times 10^{-2}$ | | Fang Lee (2007) | Q | 721 |
| | $5.0 \times 10^{-2}$ | | Fang Lee (2007) | Q | 722 |
| | $4.1 \times 10^{-2}$ | | Dunnivant et al. (1992) | Q | |
| | $4.5 \times 10^{-2}$ | | Sabljić and Güsten (1989) | Q | |
| | $6.2 \times 10^{-2}$ | | Duchowicz et al. (2020) | ? | 185, 21 |
| 2,3',5'-trichlorobiphenyl | $1.3 \times 10^{-2}$ | 5800 | Paasivirta and Sinkkonen (2009) | V | |
| $C_{12}H_7Cl_3$ | $3.4 \times 10^{-2}$ | | Burkhard et al. (1985) | V | |
| (PCB-34) | $4.5 \times 10^{-2}$ | | Keshavarz et al. (2022) | Q | |
| [37680-68-5] | $6.0 \times 10^{-2}$ | | Duchowicz et al. (2020) | Q | 184 |
| GXVMAQACUOSFJF-UHFFFAOYSA-N | $7.3 \times 10^{-2}$ | | Hilal et al. (2008) | Q | |
| | $7.1 \times 10^{-2}$ | | Modarresi et al. (2007) | Q | 67 |
| | $1.1 \times 10^{-1}$ | | Fang Lee (2007) | Q | 721 |
| | $3.9 \times 10^{-2}$ | | Fang Lee (2007) | Q | 722 |
| | $2.3 \times 10^{-2}$ | | Dunnivant et al. (1992) | Q | |
| | $2.0 \times 10^{-2}$ | | Sabljić and Güsten (1989) | Q | |
| | $4.9 \times 10^{-2}$ | | Duchowicz et al. (2020) | ? | 185, 21 |





Table A6.3: Polychlorinated biphenyls (PCBs) (. . . continued)

| Substance Formula (Trivial Name) [CAS Registry Number] InChIKey | $H_s^{cp}$ (at $T^{\ominus}$) $\left[\dfrac{\text{mol}}{\text{m}^3\,\text{Pa}}\right]$ | $\dfrac{\text{d}\ln H_s^{cp}}{\text{d}(1/T)}$ [K] | Reference | Type | Note |
|---|---|---|---|---|---|
| 3,3',4-trichlorobiphenyl | $1.8\times10^{-2}$ | 5600 | Paasivirta and Sinkkonen (2009) | V | |
| $C_{12}H_7Cl_3$ | $1.2\times10^{-1}$ | | Burkhard et al. (1985) | V | |
| (PCB-35) | $9.5\times10^{-2}$ | | Fang Lee (2007) | Q | 721 |
| [37680-69-6] | $5.9\times10^{-2}$ | | Fang Lee (2007) | Q | 722 |
| JHBVPKZLIBDTJR-UHFFFAOYSA-N | $5.5\times10^{-2}$ | | Dunnivant et al. (1992) | Q | |
| | $4.4\times10^{-2}$ | | Sabljić and Güsten (1989) | Q | |
| 3,3',5-trichlorobiphenyl | $5.8\times10^{-2}$ | | Brunner et al. (1990) | M | |
| $C_{12}H_7Cl_3$ | $1.3\times10^{-2}$ | 5600 | Paasivirta and Sinkkonen (2009) | V | |
| (PCB-36) | $7.2\times10^{-2}$ | | Burkhard et al. (1985) | V | |
| [38444-87-0] | $4.5\times10^{-2}$ | | Keshavarz et al. (2022) | Q | |
| RIBGNAJQTOXRDK-UHFFFAOYSA-N | $6.0\times10^{-2}$ | | Duchowicz et al. (2020) | Q | 299 |
| | $1.2\times10^{-1}$ | | Hilal et al. (2008) | Q | |
| | $8.8\times10^{-2}$ | | Modarresi et al. (2007) | Q | 67 |
| | $1.3\times10^{-1}$ | | Fang Lee (2007) | Q | 721 |
| | $4.7\times10^{-2}$ | | Fang Lee (2007) | Q | 722 |
| | $5.8\times10^{-2}$ | | Yaffe et al. (2003) | Q | 248, 249 |
| | $2.9\times10^{-2}$ | | Dunnivant et al. (1992) | Q | |
| | $1.9\times10^{-2}$ | | Sabljić and Güsten (1989) | Q | |
| | $5.8\times10^{-2}$ | | Duchowicz et al. (2020) | ? | 185, 21 |
| 3,4,4'-trichlorobiphenyl | $9.9\times10^{-2}$ | | Brunner et al. (1990) | M | |
| $C_{12}H_7Cl_3$ | $6.5\times10^{-2}$ | | Murphy et al. (1987) | M | 12 |
| (PCB-37) | $1.2\times10^{-2}$ | | Atlas et al. (1982) | M | 679 |
| [38444-90-5] | $1.3\times10^{-2}$ | 5400 | Paasivirta and Sinkkonen (2009) | V | |
| YZANRISAORXTHU-UHFFFAOYSA-N | $1.2\times10^{-2}$ | | Shiu and Mackay (1986) | V | |
| | $1.4\times10^{-1}$ | | Burkhard et al. (1985) | V | |
| | $4.5\times10^{-2}$ | | Keshavarz et al. (2022) | Q | |
| | $1.0\times10^{-1}$ | | Duchowicz et al. (2020) | Q | 184 |
| | $1.8\times10^{-1}$ | | Hilal et al. (2008) | Q | |
| | $1.3\times10^{-1}$ | | Modarresi et al. (2007) | Q | 67 |
| | $4.8\times10^{-2}$ | | Fang Lee (2007) | Q | 721 |
| | $6.1\times10^{-2}$ | | Fang Lee (2007) | Q | 722 |
| | $6.5\times10^{-2}$ | | Dunnivant et al. (1992) | Q | |
| | $6.9\times10^{-2}$ | | Sabljić and Güsten (1989) | Q | |
| | $9.9\times10^{-2}$ | | Duchowicz et al. (2020) | ? | 185, 21 |
| 3,4,5-trichlorobiphenyl | $1.2\times10^{-2}$ | 5400 | Paasivirta and Sinkkonen (2009) | V | |
| $C_{12}H_7Cl_3$ | $1.3\times10^{-1}$ | | Burkhard et al. (1985) | V | |
| (PCB-38) | $8.8\times10^{-2}$ | | Fang Lee (2007) | Q | 721 |
| [53555-66-1] | $5.2\times10^{-2}$ | | Fang Lee (2007) | Q | 722 |
| BSFZSQRJGZHMMV-UHFFFAOYSA-N | $4.2\times10^{-2}$ | | Dunnivant et al. (1992) | Q | |
| | $4.8\times10^{-2}$ | | Sabljić and Güsten (1989) | Q | |



Table A6.3: Polychlorinated biphenyls (PCBs) (. . . continued)

| Substance<br>Formula<br>(Trivial Name)<br>[CAS Registry Number]<br>InChIKey | $H_s^{cp}$ (at $T^{\ominus}$) $\left[\dfrac{\text{mol}}{\text{m}^3\,\text{Pa}}\right]$ | $\dfrac{\text{d}\ln H_s^{cp}}{\text{d}(1/T)}$ [K] | Reference | Type | Note |
|---|---|---|---|---|---|
| 3,4',5-trichlorobiphenyl | $1.2\times10^{-2}$ | 5600 | Paasivirta and Sinkkonen (2009) | V | |
| $C_{12}H_7Cl_3$ | $8.0\times10^{-2}$ | | Burkhard et al. (1985) | V | |
| (PCB-39) | $6.6\times10^{-2}$ | | Fang Lee (2007) | Q | 721 |
| [38444-88-1] | $4.4\times10^{-2}$ | | Fang Lee (2007) | Q | 722 |
| SYSBNFJJSJLZMM-UHFFFAOYSA-N | $3.3\times10^{-2}$ | | Dunnivant et al. (1992) | Q | |
| | $2.3\times10^{-2}$ | | Sabljić and Güsten (1989) | Q | |
| 2,2',3,3'-tetrachlorobiphenyl | $3.6\times10^{-2}$ | 3600 | Bamford et al. (2002) | M | |
| $C_{12}H_6Cl_4$ | $9.9\times10^{-2}$ | | Brunner et al. (1990) | M | |
| (PCB-40) | $4.9\times10^{-2}$ | | Dunnivant et al. (1988) | M | |
| [38444-93-8] | $4.9\times10^{-2}$ | | Dunnivant and Elzerman (1988) | M | 723 |
| VTLYHLREPCPDKX-UHFFFAOYSA-N | $6.1\times10^{-2}$ | | Murphy et al. (1987) | M | 12 |
| | $8.2\times10^{-2}$ | | Oliver (1985) | M | |
| | $1.8\times10^{-3}$ | 5300 | Paasivirta and Sinkkonen (2009) | V | |
| | $4.6\times10^{-2}$ | | Mackay et al. (2006b) | V | |
| | $4.6\times10^{-2}$ | | Mackay et al. (1992a) | V | |
| | $4.5\times10^{-2}$ | | Shiu and Mackay (1986) | V | |
| | $4.9\times10^{-2}$ | | Burkhard et al. (1985) | V | |
| | $2.4\times10^{-1}$ | | Keshavarz et al. (2022) | Q | |
| | $1.9\times10^{-1}$ | | Duchowicz et al. (2020) | Q | 299 |
| | $1.1\times10^{-1}$ | | Hilal et al. (2008) | Q | |
| | $6.8\times10^{-2}$ | | Modarresi et al. (2007) | Q | 67 |
| | $1.2\times10^{-1}$ | | Fang Lee (2007) | Q | 721 |
| | $9.7\times10^{-2}$ | | Fang Lee (2007) | Q | 722 |
| | $2.3\times10^{-1}$ | | Yaffe et al. (2003) | Q | 248, 272 |
| | $6.5\times10^{-2}$ | | English and Carroll (2001) | Q | 230, 231 |
| | $5.4\times10^{-2}$ | | Dunnivant et al. (1992) | Q | |
| | $9.9\times10^{-2}$ | | Duchowicz et al. (2020) | ? | 185, 21 |
| 2,2',3,4-tetrachlorobiphenyl | $4.9\times10^{-2}$ | | Murphy et al. (1987) | M | 12 |
| $C_{12}H_6Cl_4$ | $1.6\times10^{-2}$ | 6200 | Paasivirta and Sinkkonen (2009) | V | |
| (PCB-41) | $4.2\times10^{-2}$ | | Burkhard et al. (1985) | V | |
| [52663-59-9] | $2.4\times10^{-1}$ | | Keshavarz et al. (2022) | Q | |
| SEWHDNLIHDBVDZ-UHFFFAOYSA-N | $1.4\times10^{-1}$ | | Duchowicz et al. (2020) | Q | 184 |
| | $6.9\times10^{-2}$ | | Hilal et al. (2008) | Q | |
| | $8.7\times10^{-2}$ | | Modarresi et al. (2007) | Q | 67 |
| | $7.9\times10^{-2}$ | | Fang Lee (2007) | Q | 721 |
| | $7.1\times10^{-2}$ | | Fang Lee (2007) | Q | 722 |
| | $4.0\times10^{-2}$ | | Dunnivant et al. (1992) | Q | |
| | $4.8\times10^{-2}$ | | Sabljić and Güsten (1989) | Q | |
| | $7.0\times10^{-2}$ | | Duchowicz et al. (2020) | ? | 185, 21 |
| 2,2',3,4'-tetrachlorobiphenyl | $2.8\times10^{-2}$ | 3100 | Bamford et al. (2002) | M | |
| $C_{12}H_6Cl_4$ | $5.0\times10^{-2}$ | | Murphy et al. (1987) | M | 12 |
| (PCB-42) | $8.6\times10^{-3}$ | 5900 | Paasivirta and Sinkkonen (2009) | V | |
| [36559-22-5] | $3.4\times10^{-2}$ | | Burkhard et al. (1985) | V | |
| ALFHIHDQSYXSGP-UHFFFAOYSA-N | $2.4\times10^{-1}$ | | Keshavarz et al. (2022) | Q | |
| | $1.1\times10^{-1}$ | | Duchowicz et al. (2020) | Q | 299 |
| | $7.2\times10^{-2}$ | | Hilal et al. (2008) | Q | |





Table A6.3: Polychlorinated biphenyls (PCBs) (. . . continued)

| Substance<br>Formula<br>(Trivial Name)<br>[CAS Registry Number]<br>InChIKey | $H_s^{cp}$<br>(at $T^{\ominus}$)<br>$\left[\dfrac{\mathrm{mol}}{\mathrm{m^3\,Pa}}\right]$ | $\dfrac{\mathrm{d}\ln H_s^{cp}}{\mathrm{d}(1/T)}$<br><br>[K] | Reference | Type | Note |
|---|---|---|---|---|---|
| | $8.1\times10^{-2}$ | | Modarresi et al. (2007) | Q | 67 |
| | $6.0\times10^{-2}$ | | Fang Lee (2007) | Q | 721 |
| | $6.7\times10^{-2}$ | | Fang Lee (2007) | Q | 722 |
| | $3.9\times10^{-2}$ | | Dunnivant et al. (1992) | Q | |
| | $3.1\times10^{-2}$ | | Sabljić and Güsten (1989) | Q | |
| | $7.0\times10^{-2}$ | | Duchowicz et al. (2020) | ? | 185, 21 |
| 2,2',3,5-tetrachlorobiphenyl | $1.3\times10^{-2}$ | 6300 | Paasivirta and Sinkkonen (2009) | V | |
| $C_{12}H_6Cl_4$ | $2.4\times10^{-2}$ | | Burkhard et al. (1985) | V | |
| (PCB-43) | $1.1\times10^{-1}$ | | Fang Lee (2007) | Q | 721 |
| [70362-46-8] | $6.1\times10^{-2}$ | | Fang Lee (2007) | Q | 722 |
| NRBNBYFPJCCKTO-UHFFFAOYSA-N | $2.9\times10^{-2}$ | | Dunnivant et al. (1992) | Q | |
| | $2.8\times10^{-2}$ | | Sabljić and Güsten (1989) | Q | |
| 2,2',3,5'-tetrachlorobiphenyl | $5.4\times10^{-3}$ | | Bhangare et al. (2019) | M | 725 |
| $C_{12}H_6Cl_4$ | $3.4\times10^{-2}$ | | Bhangare et al. (2019) | M | 726 |
| (PCB-44) | $3.6\times10^{-2}$ | 3100 | Bamford et al. (2000) | M | |
| [41464-39-5] | $5.2\times10^{-2}$ | | Murphy et al. (1987) | M | 12 |
| ALDJIKXAHSDLLB-UHFFFAOYSA-N | $4.1\times10^{-2}$ | | Murphy et al. (1983a) | M | 24 |
| | $1.3\times10^{-2}$ | | Atlas et al. (1982) | M | 679 |
| | $1.1\times10^{-2}$ | 6000 | Paasivirta and Sinkkonen (2009) | V | |
| | $2.0\times10^{-2}$ | | Shiu and Mackay (1986) | V | |
| | $3.0\times10^{-2}$ | | Burkhard et al. (1985) | V | |
| | $2.4\times10^{-1}$ | | Keshavarz et al. (2022) | Q | |
| | $1.0\times10^{-1}$ | | Duchowicz et al. (2020) | Q | 184 |
| | $9.7\times10^{-2}$ | | Hilal et al. (2008) | Q | |
| | $7.7\times10^{-2}$ | | Modarresi et al. (2007) | Q | 67 |
| | $1.2\times10^{-1}$ | | Fang Lee (2007) | Q | 721 |
| | $6.7\times10^{-2}$ | | Fang Lee (2007) | Q | 722 |
| | | 4600 | Kühne et al. (2005) | Q | |
| | $4.3\times10^{-2}$ | | Dunnivant et al. (1992) | Q | |
| | $3.9\times10^{-2}$ | | Sabljić and Güsten (1989) | Q | |
| | $7.0\times10^{-2}$ | | Duchowicz et al. (2020) | ? | 185, 21 |
| | | 3400 | Kühne et al. (2005) | ? | |
| 2,2',3,6-tetrachlorobiphenyl | $2.5\times10^{-2}$ | 2900 | Bamford et al. (2002) | M | |
| $C_{12}H_6Cl_4$ | $3.8\times10^{-3}$ | 6000 | Paasivirta and Sinkkonen (2009) | V | |
| (PCB-45) | $9.9\times10^{-3}$ | | Burkhard et al. (1985) | V | |
| [70362-45-7] | $6.7\times10^{-2}$ | | Fang Lee (2007) | Q | 721 |
| VHGHHZZTMJLTJX-UHFFFAOYSA-N | $5.1\times10^{-2}$ | | Fang Lee (2007) | Q | 722 |
| | $2.8\times10^{-2}$ | | Dunnivant et al. (1992) | Q | |
| | $3.1\times10^{-2}$ | | Sabljić and Güsten (1989) | Q | |



Table A6.3: Polychlorinated biphenyls (PCBs) (. . . continued)

| Substance Formula (Trivial Name) [CAS Registry Number] InChIKey | $H_s^{cp}$ (at $T^{\ominus}$) $\left[\dfrac{\text{mol}}{\text{m}^3\,\text{Pa}}\right]$ | $\dfrac{\text{d}\ln H_s^{cp}}{\text{d}(1/T)}$ [K] | Reference | Type | Note |
|---|---|---|---|---|---|
| 2,2',3,6'-tetrachlorobiphenyl | $3.0\times10^{-2}$ | 3400 | Bamford et al. (2002) | M | |
| $C_{12}H_6Cl_4$ | $3.8\times10^{-2}$ | | Murphy et al. (1987) | M | 12 |
| (PCB-46) | $9.1\times10^{-4}$ | 5300 | Paasivirta and Sinkkonen (2009) | V | |
| [41464-47-5] | $1.4\times10^{-2}$ | | Burkhard et al. (1985) | V | |
| CUGLICQCTXWQNF-UHFFFAOYSA-N | $6.7\times10^{-2}$ | | Fang Lee (2007) | Q | 721 |
| | $5.7\times10^{-2}$ | | Fang Lee (2007) | Q | 722 |
| | $2.9\times10^{-2}$ | | Dunnivant et al. (1992) | Q | |
| | $2.7\times10^{-2}$ | | Sabljić and Güsten (1989) | Q | |
| 2,2',4,4'-tetrachlorobiphenyl | $2.1\times10^{-1}$ | | Lau et al. (2006) | M | 719 |
| $C_{12}H_6Cl_4$ | $9.1\times10^{-3}$ | | Lau et al. (2006) | M | 720 |
| (PCB-47) | $1.8\times10^{-1}$ | -6000 | Charles and Destaillats (2005) | M | |
| [2437-79-8] | $5.2\times10^{-2}$ | | Brunner et al. (1990) | M | |
| QORAVNMWUNPXAO-UHFFFAOYSA-N | $2.0\times10^{-2}$ | 6200 | Paasivirta and Sinkkonen (2009) | V | |
| | $5.7\times10^{-2}$ | | Mackay et al. (2006b) | V | |
| | $5.8\times10^{-2}$ | | Mackay et al. (1992a) | V | |
| | $2.0\times10^{-3}$ | | Hwang et al. (1992) | V | |
| | $5.9\times10^{-2}$ | | Shiu and Mackay (1986) | V | |
| | $2.3\times10^{-2}$ | | Burkhard et al. (1985) | V | |
| | $2.4\times10^{-1}$ | | Keshavarz et al. (2022) | Q | |
| | $6.6\times10^{-2}$ | | Duchowicz et al. (2020) | Q | 299 |
| | $5.0\times10^{-2}$ | | Hilal et al. (2008) | Q | |
| | $8.2\times10^{-2}$ | | Modarresi et al. (2007) | Q | 67 |
| | $3.5\times10^{-2}$ | | Fang Lee (2007) | Q | 721 |
| | $4.8\times10^{-2}$ | | Fang Lee (2007) | Q | 722 |
| | $2.7\times10^{-2}$ | | Dunnivant et al. (1992) | Q | |
| | $2.2\times10^{-2}$ | | Sabljić and Güsten (1989) | Q | |
| | $5.2\times10^{-2}$ | | Duchowicz et al. (2020) | ? | 185, 21 |
| 2,2',4,5-tetrachlorobiphenyl | $2.7\times10^{-2}$ | 3000 | Bamford et al. (2002) | M | |
| $C_{12}H_6Cl_4$ | $3.9\times10^{-2}$ | | Murphy et al. (1987) | M | 12 |
| (PCB-48) | $6.1\times10^{-3}$ | 6100 | Paasivirta and Sinkkonen (2009) | V | |
| [70362-47-9] | $2.5\times10^{-2}$ | | Burkhard et al. (1985) | V | |
| XBTHILIDLBPRPM-UHFFFAOYSA-N | $6.3\times10^{-2}$ | | Fang Lee (2007) | Q | 721 |
| | $5.3\times10^{-2}$ | | Fang Lee (2007) | Q | 722 |
| | $3.3\times10^{-2}$ | | Dunnivant et al. (1992) | Q | |
| | $3.8\times10^{-2}$ | | Sabljić and Güsten (1989) | Q | |
| 2,2',4,5'-tetrachlorobiphenyl | $2.7\times10^{-2}$ | 3000 | Bamford et al. (2002) | M | |
| $C_{12}H_6Cl_4$ | $3.6\times10^{-2}$ | | Murphy et al. (1987) | M | 12 |
| (PCB-49) | $4.9\times10^{-2}$ | | Murphy et al. (1983a) | M | 24 |
| [41464-40-8] | $8.3\times10^{-3}$ | 5900 | Paasivirta and Sinkkonen (2009) | V | |
| ZWPVHELAQPIZHO-UHFFFAOYSA-N | $5.0\times10^{-2}$ | | Shiu and Mackay (1986) | V | |
| | $2.1\times10^{-2}$ | | Burkhard et al. (1985) | V | |
| | $2.4\times10^{-1}$ | | Keshavarz et al. (2022) | Q | |
| | $6.1\times10^{-2}$ | | Duchowicz et al. (2020) | Q | 299 |
| | $6.1\times10^{-2}$ | | Hilal et al. (2008) | Q | |
| | $7.6\times10^{-2}$ | | Modarresi et al. (2007) | Q | 67 |
| | $7.0\times10^{-2}$ | | Fang Lee (2007) | Q | 721 |



Table A6.3: Polychlorinated biphenyls (PCBs) (...continued)

| Substance Formula (Trivial Name) [CAS Registry Number] InChIKey | $H_s^{cp}$ (at $T^\ominus$) $\left[\dfrac{\text{mol}}{\text{m}^3\,\text{Pa}}\right]$ | $\dfrac{\text{d}\ln H_s^{cp}}{\text{d}(1/T)}$ [K] | Reference | Type | Note |
|---|---|---|---|---|---|
| | $4.5\times10^{-2}$ | | Fang Lee (2007) | Q | 722 |
| | $2.8\times10^{-2}$ | | Dunnivant et al. (1992) | Q | |
| | $2.6\times10^{-2}$ | | Sabljić and Güsten (1989) | Q | |
| | $4.7\times10^{-2}$ | | Duchowicz et al. (2020) | ? | 185, 21 |
| 2,2',4,6-tetrachlorobiphenyl $C_{12}H_6Cl_4$ (PCB-50) [62796-65-0] VLLVVZDKBSYMCG-UHFFFAOYSA-N | $1.6\times10^{-2}$ | 2900 | Bamford et al. (2000) | M | |
| | $1.3\times10^{-2}$ | | Atlas et al. (1982) | M | 679 |
| | $9.9\times10^{-3}$ | 6300 | Paasivirta and Sinkkonen (2009) | V | |
| | $1.3\times10^{-2}$ | | Shiu and Mackay (1986) | V | |
| | $7.3\times10^{-3}$ | | Burkhard et al. (1985) | V | |
| | $3.1\times10^{-2}$ | | Bhangare et al. (2019) | Q | |
| | $3.9\times10^{-2}$ | | Fang Lee (2007) | Q | 721 |
| | $2.8\times10^{-2}$ | | Fang Lee (2007) | Q | 722 |
| | | 3600 | Kühne et al. (2005) | Q | |
| | $1.6\times10^{-2}$ | | Dunnivant et al. (1992) | Q | |
| | $1.7\times10^{-2}$ | | Sabljić and Güsten (1989) | Q | |
| | | 3100 | Kühne et al. (2005) | ? | |
| 2,2',4,6'-tetrachlorobiphenyl $C_{12}H_6Cl_4$ (PCB-51) [68194-04-7] WVHNUGRFECMVLQ-UHFFFAOYSA-N | $2.5\times10^{-2}$ | 6300 | Paasivirta and Sinkkonen (2009) | V | |
| | $9.9\times10^{-3}$ | | Burkhard et al. (1985) | V | |
| | $2.4\times10^{-1}$ | | Keshavarz et al. (2022) | Q | |
| | $6.6\times10^{-2}$ | | Duchowicz et al. (2020) | Q | 184 |
| | $7.3\times10^{-2}$ | | Hilal et al. (2008) | Q | |
| | $5.9\times10^{-2}$ | | Modarresi et al. (2007) | Q | 67 |
| | $3.8\times10^{-2}$ | | Fang Lee (2007) | Q | 721 |
| | $4.0\times10^{-2}$ | | Fang Lee (2007) | Q | 722 |
| | $1.9\times10^{-2}$ | | Dunnivant et al. (1992) | Q | |
| | $2.0\times10^{-2}$ | | Sabljić and Güsten (1989) | Q | |
| | $7.0\times10^{-2}$ | | Duchowicz et al. (2020) | ? | 185, 21 |
| 2,2',5,5'-tetrachlorobiphenyl $C_{12}H_6Cl_4$ (PCB-52) [35693-99-3] HCWZEPKLWVAEOV-UHFFFAOYSA-N | $3.5\times10^{-2}$ | 6600 | Li et al. (2003) | L | 366 |
| | $4.0\times10^{-2}$ | 6800 | Li et al. (2003) | L | 367 |
| | $4.9\times10^{-3}$ | | Bhangare et al. (2019) | M | 725 |
| | $2.9\times10^{-2}$ | | Bhangare et al. (2019) | M | 726 |
| | $3.2\times10^{-2}$ | 3700 | Bamford et al. (2000) | M | |
| | $4.2\times10^{-2}$ | 6200 | ten Hulscher et al. (1992) | M | |
| | $4.9\times10^{-2}$ | | Brunner et al. (1990) | M | |
| | $2.9\times10^{-2}$ | | Dunnivant et al. (1988) | M | |
| | $2.9\times10^{-2}$ | | Dunnivant and Elzerman (1988) | M | 723 |
| | $4.1\times10^{-2}$ | | Murphy et al. (1987) | M | 12 |
| | $8.2\times10^{-2}$ | | Oliver (1985) | M | |
| | $4.5\times10^{-2}$ | | Murphy et al. (1983a) | M | 24 |
| | $1.1\times10^{-2}$ | | Atlas et al. (1982) | M | 679 |
| | | | Westcott et al. (1981) | M | 727 |
| | $3.7\times10^{-3}$ | 5700 | Paasivirta and Sinkkonen (2009) | V | |
| | $2.1\times10^{-2}$ | | Mackay et al. (2006b) | V | |
| | $2.1\times10^{-2}$ | | Mackay et al. (1992a) | V | |
| | $1.2\times10^{-1}$ | | McLachlan et al. (1990) | V | 373 |
| | $2.1\times10^{-2}$ | | Shiu and Mackay (1986) | V | |



Table A6.3: Polychlorinated biphenyls (PCBs) (...continued)

| Substance Formula (Trivial Name) [CAS Registry Number] InChIKey | $H_s^{cp}$ (at $T^{\ominus}$) $\left[\dfrac{\text{mol}}{\text{m}^3\,\text{Pa}}\right]$ | $\dfrac{\text{d}\ln H_s^{cp}}{\text{d}(1/T)}$ [K] | Reference | Type | Note |
|---|---|---|---|---|---|
| | $1.9\times10^{-2}$ | | Burkhard et al. (1985) | V | |
| | $1.2\times10^{-2}$ | 7700 | Paasivirta et al. (1999) | T | |
| | $3.8\times10^{-2}$ | | Murphy et al. (1983b) | X | 724, 24 |
| | $2.4\times10^{-1}$ | | Keshavarz et al. (2022) | Q | |
| | $5.6\times10^{-2}$ | | Duchowicz et al. (2020) | Q | 184 |
| | $7.9\times10^{-2}$ | | Hilal et al. (2008) | Q | |
| | $6.5\times10^{-2}$ | | Modarresi et al. (2007) | Q | 67 |
| | $9.7\times10^{-2}$ | | Fang Lee (2007) | Q | 721 |
| | $4.6\times10^{-2}$ | | Fang Lee (2007) | Q | 722 |
| | | 4200 | Kühne et al. (2005) | Q | |
| | $3.1\times10^{-2}$ | | Dunnivant et al. (1992) | Q | |
| | $4.9\times10^{-2}$ | | Duchowicz et al. (2020) | ? | 185, 21 |
| | | 4900 | Kühne et al. (2005) | ? | |
| 2,2',5,6'-tetrachlorobiphenyl $C_{12}H_6Cl_4$ (PCB-53) [41464-41-9] SFTUSTXGTCCSHX-UHFFFAOYSA-N | $2.4\times10^{-2}$ | | Dunnivant et al. (1988) | M | |
| | $2.4\times10^{-2}$ | | Dunnivant and Elzerman (1988) | M | 723 |
| | $3.5\times10^{-2}$ | | Murphy et al. (1987) | M | 12 |
| | $3.3\times10^{-2}$ | | Murphy et al. (1983a) | M | 24 |
| | $5.4\times10^{-2}$ | | Duchowicz et al. (2020) | V | 186 |
| | $1.6\times10^{-3}$ | 5500 | Paasivirta and Sinkkonen (2009) | V | |
| | $3.3\times10^{-2}$ | | Shiu and Mackay (1986) | V | |
| | $8.9\times10^{-3}$ | | Burkhard et al. (1985) | V | |
| | $6.1\times10^{-2}$ | | Duchowicz et al. (2020) | Q | |
| | $8.8\times10^{-2}$ | | Hilal et al. (2008) | Q | |
| | $5.9\times10^{-2}$ | | Modarresi et al. (2007) | Q | 67 |
| | $5.3\times10^{-2}$ | | Fang Lee (2007) | Q | 721 |
| | $4.1\times10^{-2}$ | | Fang Lee (2007) | Q | 722 |
| | $2.3\times10^{-2}$ | | Dunnivant et al. (1992) | Q | |
| 2,2',6,6'-tetrachlorobiphenyl $C_{12}H_6Cl_4$ (PCB-54) [15968-05-5] PXAGFNRKXSYIHU-UHFFFAOYSA-N | $4.9\times10^{-2}$ | | Brunner et al. (1990) | M | |
| | $1.8\times10^{-2}$ | | Dunnivant et al. (1988) | M | |
| | $1.8\times10^{-2}$ | | Dunnivant and Elzerman (1988) | M | 723 |
| | $1.0\times10^{-4}$ | 4800 | Paasivirta and Sinkkonen (2009) | V | |
| | $5.3\times10^{-3}$ | | Burkhard et al. (1985) | V | |
| | $6.7\times10^{-2}$ | | Dunnivant et al. (1988) | C | |
| | $2.4\times10^{-1}$ | | Keshavarz et al. (2022) | Q | |
| | $6.6\times10^{-2}$ | | Duchowicz et al. (2020) | Q | 184 |
| | $1.2\times10^{-1}$ | | Hilal et al. (2008) | Q | |
| | $6.5\times10^{-2}$ | | Modarresi et al. (2007) | Q | 67 |
| | $3.3\times10^{-2}$ | | Fang Lee (2007) | Q | 721 |
| | $2.7\times10^{-2}$ | | Fang Lee (2007) | Q | 722 |
| | $5.2\times10^{-2}$ | | Yaffe et al. (2003) | Q | 248, 249 |
| | $1.7\times10^{-2}$ | | Dunnivant et al. (1992) | Q | |
| | $4.9\times10^{-2}$ | | Duchowicz et al. (2020) | ? | 185, 21 |



Table A6.3: Polychlorinated biphenyls (PCBs) (...continued)

| Substance Formula (Trivial Name) [CAS Registry Number] InChIKey | $H_s^{cp}$ (at $T^\ominus$) $\left[\dfrac{\mathrm{mol}}{\mathrm{m}^3\,\mathrm{Pa}}\right]$ | $\dfrac{\mathrm{d}\ln H_s^{cp}}{\mathrm{d}(1/T)}$ [K] | Reference | Type | Note |
|---|---|---|---|---|---|
| 2,3,3',4-tetrachlorobiphenyl | $9.6\times10^{-3}$ | 5900 | Paasivirta and Sinkkonen (2009) | V | |
| $C_{12}H_6Cl_4$ | $8.7\times10^{-2}$ | | Burkhard et al. (1985) | V | |
| (PCB-55) | $9.9\times10^{-2}$ | | Fang Lee (2007) | Q | 721 |
| [74338-24-2] | $9.3\times10^{-2}$ | | Fang Lee (2007) | Q | 722 |
| ZKGSEEWIVLAUNH-UHFFFAOYSA-N | $5.4\times10^{-2}$ | | Dunnivant et al. (1992) | Q | |
| | $4.3\times10^{-2}$ | | Sabljić and Güsten (1989) | Q | |
| 2,3,3',4'-tetrachlorobiphenyl | $3.8\times10^{-2}$ | 3800 | Bamford et al. (2002) | M | |
| $C_{12}H_6Cl_4$ | $6.1\times10^{-2}$ | | Murphy et al. (1987) | M | 12 |
| (PCB-56) | $4.9\times10^{-3}$ | 5400 | Paasivirta and Sinkkonen (2009) | V | |
| [41464-43-1] | $1.1\times10^{-1}$ | | Burkhard et al. (1985) | V | |
| UNCGJRRROFURDV-UHFFFAOYSA-N | $7.5\times10^{-2}$ | | Fang Lee (2007) | Q | 721 |
| | $9.7\times10^{-2}$ | | Fang Lee (2007) | Q | 722 |
| | $6.5\times10^{-2}$ | | Dunnivant et al. (1992) | Q | |
| | $7.5\times10^{-2}$ | | Sabljić and Güsten (1989) | Q | |
| 2,3,3',5-tetrachlorobiphenyl | $8.9\times10^{-3}$ | 6100 | Paasivirta and Sinkkonen (2009) | V | |
| $C_{12}H_6Cl_4$ | $5.1\times10^{-2}$ | | Burkhard et al. (1985) | V | |
| (PCB-57) | $1.4\times10^{-1}$ | | Fang Lee (2007) | Q | 721 |
| [70424-67-8] | $7.8\times10^{-2}$ | | Fang Lee (2007) | Q | 722 |
| DHDBTLFALXRTLB-UHFFFAOYSA-N | $3.6\times10^{-2}$ | | Dunnivant et al. (1992) | Q | |
| | $2.7\times10^{-2}$ | | Sabljić and Güsten (1989) | Q | |
| 2,3,3',5'-tetrachlorobiphenyl | $2.3\times10^{-3}$ | 5400 | Paasivirta and Sinkkonen (2009) | V | |
| $C_{12}H_6Cl_4$ | $6.2\times10^{-2}$ | | Burkhard et al. (1985) | V | |
| (PCB-58) | $1.5\times10^{-1}$ | | Fang Lee (2007) | Q | 721 |
| [41464-49-7] | $8.1\times10^{-2}$ | | Fang Lee (2007) | Q | 722 |
| IOPBNBSKOPJKEG-UHFFFAOYSA-N | $3.9\times10^{-2}$ | | Dunnivant et al. (1992) | Q | |
| | $2.4\times10^{-2}$ | | Sabljić and Güsten (1989) | Q | |
| 2,3,3',6-tetrachlorobiphenyl | $4.4\times10^{-2}$ | 6600 | Paasivirta and Sinkkonen (2009) | V | |
| $C_{12}H_6Cl_4$ | $2.9\times10^{-2}$ | | Burkhard et al. (1985) | V | |
| (PCB-59) | $8.3\times10^{-2}$ | | Fang Lee (2007) | Q | 721 |
| [74472-33-6] | $8.3\times10^{-2}$ | | Fang Lee (2007) | Q | 722 |
| WZNAMGYIQPAXDH-UHFFFAOYSA-N | $3.2\times10^{-2}$ | | Dunnivant et al. (1992) | Q | |
| | $2.7\times10^{-2}$ | | Sabljić and Güsten (1989) | Q | |
| 2,3,4,4'-tetrachlorobiphenyl | $6.1\times10^{-2}$ | | Murphy et al. (1987) | M | 12 |
| $C_{12}H_6Cl_4$ | $1.2\times10^{-2}$ | | Atlas et al. (1982) | M | 679 |
| (PCB-60) | $2.9\times10^{-3}$ | 5500 | Paasivirta and Sinkkonen (2009) | V | |
| [33025-41-1] | $9.7\times10^{-2}$ | | Burkhard et al. (1985) | V | |
| XLDBTRJKXLKYTC-UHFFFAOYSA-N | $4.9\times10^{-2}$ | | Fang Lee (2007) | Q | 721 |
| | $9.2\times10^{-2}$ | | Fang Lee (2007) | Q | 722 |
| | $6.5\times10^{-2}$ | | Dunnivant et al. (1992) | Q | |
| | $6.5\times10^{-2}$ | | Sabljić and Güsten (1989) | Q | |





Table A6.3: Polychlorinated biphenyls (PCBs) (...continued)

| Substance<br>Formula<br>(Trivial Name)<br>[CAS Registry Number]<br>InChIKey | $H_s^{cp}$<br>(at $T^{\ominus}$)<br>$\left[\dfrac{\mathrm{mol}}{\mathrm{m^3\,Pa}}\right]$ | $\dfrac{\mathrm{d}\ln H_s^{cp}}{\mathrm{d}(1/T)}$<br><br>[K] | Reference | Type | Note |
|---|---|---|---|---|---|
| 2,3,4,5-tetrachlorobiphenyl | $4.9\times10^{-2}$ | 6600 | Li et al. (2003) | L | 366 |
| $C_{12}H_6Cl_4$ | $5.0\times10^{-2}$ | 7200 | Li et al. (2003) | L | 367 |
| (PCB-61) | $9.6\times10^{-3}$ | | Duchowicz et al. (2020) | V | 186 |
| [33284-53-6] | $4.9\times10^{-3}$ | 5600 | Paasivirta and Sinkkonen (2009) | V | |
| HLQDGCWIOSOMDP-UHFFFAOYSA-N | $8.7\times10^{-2}$ | | Burkhard et al. (1985) | V | |
| | $9.6\times10^{-2}$ | | Duchowicz et al. (2020) | Q | |
| | $9.0\times10^{-2}$ | | Fang Lee (2007) | Q | 721 |
| | $8.4\times10^{-2}$ | | Fang Lee (2007) | Q | 722 |
| | $4.1\times10^{-2}$ | | Dunnivant et al. (1992) | Q | |
| | $5.7\times10^{-2}$ | | Sabljić and Güsten (1989) | Q | |
| 2,3,4,6-tetrachlorobiphenyl | $4.7\times10^{-2}$ | | Brunner et al. (1990) | M | |
| $C_{12}H_6Cl_4$ | $7.1\times10^{-3}$ | 6000 | Paasivirta and Sinkkonen (2009) | V | |
| (PCB-62) | $3.3\times10^{-2}$ | | Burkhard et al. (1985) | V | |
| [54230-22-7] | $4.5\times10^{-2}$ | | Keshavarz et al. (2022) | Q | |
| HOBRTVXSIVSXIA-UHFFFAOYSA-N | $5.7\times10^{-2}$ | | Duchowicz et al. (2020) | Q | 299 |
| | $4.0\times10^{-2}$ | | Hilal et al. (2008) | Q | |
| | $1.8\times10^{-1}$ | | Modarresi et al. (2007) | Q | 67 |
| | $5.5\times10^{-2}$ | | Fang Lee (2007) | Q | 721 |
| | $6.4\times10^{-2}$ | | Fang Lee (2007) | Q | 722 |
| | $2.7\times10^{-2}$ | | Dunnivant et al. (1992) | Q | |
| | $2.7\times10^{-2}$ | | Sabljić and Güsten (1989) | Q | |
| | $4.7\times10^{-2}$ | | Duchowicz et al. (2020) | ? | 185, 21 |
| 2,3,4',5-tetrachlorobiphenyl | $2.5\times10^{-2}$ | 3000 | Bamford et al. (2002) | M | |
| $C_{12}H_6Cl_4$ | $3.4\times10^{-2}$ | | Murphy et al. (1987) | M | 12 |
| (PCB-63) | $9.4\times10^{-3}$ | 6100 | Paasivirta and Sinkkonen (2009) | V | |
| [74472-34-7] | $5.6\times10^{-2}$ | | Burkhard et al. (1985) | V | |
| CITMYAMXIZQCJD-UHFFFAOYSA-N | $6.8\times10^{-2}$ | | Fang Lee (2007) | Q | 721 |
| | $7.1\times10^{-2}$ | | Fang Lee (2007) | Q | 722 |
| | $4.1\times10^{-2}$ | | Dunnivant et al. (1992) | Q | |
| | $3.4\times10^{-2}$ | | Sabljić and Güsten (1989) | Q | |
| 2,3,4',6-tetrachlorobiphenyl | $2.5\times10^{-2}$ | 2900 | Bamford et al. (2002) | M | |
| $C_{12}H_6Cl_4$ | $5.8\times10^{-2}$ | | Murphy et al. (1987) | M | 12 |
| (PCB-64) | $7.9\times10^{-3}$ | 6000 | Paasivirta and Sinkkonen (2009) | V | |
| [52663-58-8] | $3.2\times10^{-2}$ | | Burkhard et al. (1985) | V | |
| FXRXQYZZALWWGA-UHFFFAOYSA-N | $4.5\times10^{-2}$ | | Keshavarz et al. (2022) | Q | |
| | $7.7\times10^{-2}$ | | Duchowicz et al. (2020) | Q | 299 |
| | $1.1\times10^{-1}$ | | Hilal et al. (2008) | Q | |
| | $1.3\times10^{-1}$ | | Modarresi et al. (2007) | Q | 67 |
| | $4.2\times10^{-2}$ | | Fang Lee (2007) | Q | 721 |
| | $7.7\times10^{-2}$ | | Fang Lee (2007) | Q | 722 |
| | $3.6\times10^{-2}$ | | Dunnivant et al. (1992) | Q | |
| | $3.5\times10^{-2}$ | | Sabljić and Güsten (1989) | Q | |
| | $7.0\times10^{-2}$ | | Duchowicz et al. (2020) | ? | 185, 21 |



Table A6.3: Polychlorinated biphenyls (PCBs) (...continued)

| Substance Formula (Trivial Name) [CAS Registry Number] InChIKey | $H_s^{cp}$ (at $T^{\ominus}$) $\left[\dfrac{\mathrm{mol}}{\mathrm{m^3\,Pa}}\right]$ | $\dfrac{\mathrm{d}\ln H_s^{cp}}{\mathrm{d}(1/T)}$ [K] | Reference | Type | Note |
|---|---|---|---|---|---|
| 2,3,5,6-tetrachlorobiphenyl | $4.9\times10^{-3}$ | 5800 | Paasivirta and Sinkkonen (2009) | V | |
| $C_{12}H_6Cl_4$ | $3.7\times10^{-2}$ | | Burkhard et al. (1985) | V | |
| (PCB-65) | $5.3\times10^{-2}$ | | Hilal et al. (2008) | Q | |
| [33284-54-7] | $1.4\times10^{-1}$ | | Modarresi et al. (2007) | Q | 67 |
| BLAYIQLVUNIICD-UHFFFAOYSA-N | $7.6\times10^{-2}$ | | Fang Lee (2007) | Q | 721 |
| | $9.9\times10^{-2}$ | | Fang Lee (2007) | Q | 722 |
| | $2.9\times10^{-2}$ | | Dunnivant et al. (1992) | Q | |
| | $3.2\times10^{-2}$ | | Sabljić and Güsten (1989) | Q | |
| 2,3',4,4'-tetrachlorobiphenyl | $2.7\times10^{-2}$ | 3500 | Bamford et al. (2000) | M | |
| $C_{12}H_6Cl_4$ | $4.9\times10^{-2}$ | | Murphy et al. (1987) | M | 12 |
| (PCB-66) | $3.0\times10^{-3}$ | 5300 | Paasivirta and Sinkkonen (2009) | V | |
| [32598-10-0] | $1.2\times10^{-2}$ | | Shiu and Mackay (1986) | V | |
| RKLLTEAEZIJBAU-UHFFFAOYSA-N | $7.3\times10^{-2}$ | | Burkhard et al. (1985) | V | |
| | $2.4\times10^{-1}$ | | Keshavarz et al. (2022) | Q | |
| | $1.1\times10^{-1}$ | | Duchowicz et al. (2020) | Q | 299 |
| | $2.9\times10^{-2}$ | | Bhangare et al. (2019) | Q | |
| | $2.1\times10^{-1}$ | | Hilal et al. (2008) | Q | |
| | $8.5\times10^{-2}$ | | Modarresi et al. (2007) | Q | 67 |
| | $4.3\times10^{-2}$ | | Fang Lee (2007) | Q | 721 |
| | $6.8\times10^{-2}$ | | Fang Lee (2007) | Q | 722 |
| | | 5200 | Kühne et al. (2005) | Q | |
| | $4.9\times10^{-2}$ | | Dunnivant et al. (1992) | Q | |
| | $3.9\times10^{-2}$ | | Sabljić and Güsten (1989) | Q | |
| | $8.2\times10^{-2}$ | | Duchowicz et al. (2020) | ? | 185, 21 |
| | | 3800 | Kühne et al. (2005) | ? | |
| 2,3',4,5-tetrachlorobiphenyl | $9.9\times10^{-2}$ | | Brunner et al. (1990) | M | |
| $C_{12}H_6Cl_4$ | $1.6\times10^{-2}$ | 6200 | Paasivirta and Sinkkonen (2009) | V | |
| (PCB-67) | $5.2\times10^{-2}$ | | Burkhard et al. (1985) | V | |
| [73575-53-8] | $4.5\times10^{-2}$ | | Keshavarz et al. (2022) | Q | |
| LQEGJNOKOZHBBZ-UHFFFAOYSA-N | $7.7\times10^{-2}$ | | Duchowicz et al. (2020) | Q | 299 |
| | $1.9\times10^{-1}$ | | Hilal et al. (2008) | Q | |
| | $1.2\times10^{-1}$ | | Modarresi et al. (2007) | Q | 67 |
| | $7.9\times10^{-2}$ | | Fang Lee (2007) | Q | 721 |
| | $6.8\times10^{-2}$ | | Fang Lee (2007) | Q | 722 |
| | $4.2\times10^{-2}$ | | Dunnivant et al. (1992) | Q | |
| | $3.4\times10^{-2}$ | | Sabljić and Güsten (1989) | Q | |
| | $9.9\times10^{-2}$ | | Duchowicz et al. (2020) | ? | 185, 21 |
| 2,3',4,5'-tetrachlorobiphenyl | $7.2\times10^{-3}$ | 5800 | Paasivirta and Sinkkonen (2009) | V | |
| $C_{12}H_6Cl_4$ | $4.3\times10^{-2}$ | | Burkhard et al. (1985) | V | |
| (PCB-68) | $8.7\times10^{-2}$ | | Fang Lee (2007) | Q | 721 |
| [73575-52-7] | $5.2\times10^{-2}$ | | Fang Lee (2007) | Q | 722 |
| KTTXLLZIBIDUCR-UHFFFAOYSA-N | $2.6\times10^{-2}$ | | Dunnivant et al. (1992) | Q | |
| | $1.8\times10^{-2}$ | | Sabljić and Güsten (1989) | Q | |



Table A6.3: Polychlorinated biphenyls (PCBs) (. . . continued)

| Substance Formula (Trivial Name) [CAS Registry Number] InChIKey | $H_s^{cp}$ (at $T^{\ominus}$) $\left[\dfrac{\text{mol}}{\text{m}^3\,\text{Pa}}\right]$ | $\dfrac{\text{d}\ln H_s^{cp}}{\text{d}(1/T)}$ [K] | Reference | Type | Note |
|---|---|---|---|---|---|
| 2,3',4,6-tetrachlorobiphenyl | $1.9\times10^{-2}$ | 6500 | Paasivirta and Sinkkonen (2009) | V | |
| $C_{12}H_6Cl_4$ | $2.1\times10^{-2}$ | | Burkhard et al. (1985) | V | |
| (PCB-69) | $4.5\times10^{-2}$ | | Keshavarz et al. (2022) | Q | |
| [60233-24-1] | $5.0\times10^{-2}$ | | Duchowicz et al. (2020) | Q | |
| CKUBKYSLNCKBOI-UHFFFAOYSA-N | $7.9\times10^{-2}$ | | Hilal et al. (2008) | Q | |
| | $1.2\times10^{-1}$ | | Modarresi et al. (2007) | Q | 67 |
| | $4.8\times10^{-2}$ | | Fang Lee (2007) | Q | 721 |
| | $4.5\times10^{-2}$ | | Fang Lee (2007) | Q | 722 |
| | $2.0\times10^{-2}$ | | Dunnivant et al. (1992) | Q | |
| | $1.6\times10^{-2}$ | | Sabljić and Güsten (1989) | Q | |
| | $4.7\times10^{-2}$ | | Duchowicz et al. (2020) | ? | 185, 21 |
| 2,3',4',5-tetrachlorobiphenyl | $3.3\times10^{-2}$ | 3500 | Bamford et al. (2002) | M | |
| $C_{12}H_6Cl_4$ | $9.9\times10^{-2}$ | | Brunner et al. (1990) | M | |
| (PCB-70) | $5.2\times10^{-2}$ | | Murphy et al. (1987) | M | 12 |
| [32598-11-1] | $1.1\times10^{-1}$ | | Brownawell (1986) | M | 294 |
| KENZYIHFBRWMOD-UHFFFAOYSA-N | $4.9\times10^{-2}$ | | Murphy et al. (1983a) | M | 24 |
| | $4.0\times10^{-3}$ | 5400 | Paasivirta and Sinkkonen (2009) | V | |
| | $5.0\times10^{-2}$ | | Shiu and Mackay (1986) | V | |
| | $6.5\times10^{-2}$ | | Burkhard et al. (1985) | V | |
| | $2.4\times10^{-1}$ | | Keshavarz et al. (2022) | Q | |
| | $1.0\times10^{-1}$ | | Duchowicz et al. (2020) | Q | 299 |
| | $2.0\times10^{-1}$ | | Hilal et al. (2008) | Q | |
| | $7.2\times10^{-2}$ | | Modarresi et al. (2007) | Q | 67 |
| | $6.0\times10^{-2}$ | | Fang Lee (2007) | Q | 721 |
| | $6.4\times10^{-2}$ | | Fang Lee (2007) | Q | 722 |
| | $4.9\times10^{-2}$ | | Dunnivant et al. (1992) | Q | |
| | $5.2\times10^{-2}$ | | Sabljić and Güsten (1989) | Q | |
| | $9.9\times10^{-2}$ | | Duchowicz et al. (2020) | ? | 185, 21 |
| 2,3',4',6-tetrachlorobiphenyl | $1.8\times10^{-2}$ | 6000 | Paasivirta and Sinkkonen (2009) | V | |
| $C_{12}H_6Cl_4$ | $4.4\times10^{-2}$ | | Burkhard et al. (1985) | V | |
| (PCB-71) | $3.6\times10^{-2}$ | | Fang Lee (2007) | Q | 721 |
| [41464-46-4] | $7.0\times10^{-2}$ | | Fang Lee (2007) | Q | 722 |
| WYVBETQIUHPLFO-UHFFFAOYSA-N | $3.1\times10^{-2}$ | | Dunnivant et al. (1992) | Q | |
| | $3.1\times10^{-2}$ | | Sabljić and Güsten (1989) | Q | |
| 2,3',5,5'-tetrachlorobiphenyl | $4.0\times10^{-3}$ | 5700 | Paasivirta and Sinkkonen (2009) | V | |
| $C_{12}H_6Cl_4$ | $3.9\times10^{-2}$ | | Burkhard et al. (1985) | V | |
| (PCB-72) | $1.2\times10^{-1}$ | | Fang Lee (2007) | Q | 721 |
| [41464-42-0] | $5.3\times10^{-2}$ | | Fang Lee (2007) | Q | 722 |
| WBTMFEPLVQOWFI-UHFFFAOYSA-N | $2.7\times10^{-2}$ | | Dunnivant et al. (1992) | Q | |
| | $2.1\times10^{-2}$ | | Sabljić and Güsten (1989) | Q | |



Table A6.3: Polychlorinated biphenyls (PCBs) (... continued)

| Substance<br>Formula<br>(Trivial Name)<br>[CAS Registry Number]<br>InChIKey | $H_s^{cp}$<br>(at $T^\ominus$)<br>$\left[\dfrac{\text{mol}}{\text{m}^3\,\text{Pa}}\right]$ | $\dfrac{\text{d}\ln H_s^{cp}}{\text{d}(1/T)}$<br><br>[K] | Reference | Type | Note |
|---|---|---|---|---|---|
| 2,3',5',6-tetrachlorobiphenyl | $6.4\times10^{-3}$ | 5900 | Paasivirta and Sinkkonen (2009) | V | |
| $C_{12}H_6Cl_4$ | $2.6\times10^{-2}$ | | Burkhard et al. (1985) | V | |
| (PCB-73) | $7.4\times10^{-2}$ | | Fang Lee (2007) | Q | 721 |
| [74338-23-1] | $6.0\times10^{-2}$ | | Fang Lee (2007) | Q | 722 |
| HDULUCZRGGWTMZ-UHFFFAOYSA-N | $1.9\times10^{-2}$ | | Dunnivant et al. (1992) | Q | |
| | $1.6\times10^{-2}$ | | Sabljić and Güsten (1989) | Q | |
| 2,4,4',5-tetrachlorobiphenyl | $2.6\times10^{-2}$ | 3000 | Bamford et al. (2002) | M | |
| $C_{12}H_6Cl_4$ | $9.9\times10^{-2}$ | | Brunner et al. (1990) | M | |
| (PCB-74) | $4.7\times10^{-2}$ | | Murphy et al. (1987) | M | 12 |
| [32690-93-0] | $4.8\times10^{-3}$ | 5800 | Paasivirta and Sinkkonen (2009) | V | |
| TULCXSBAPHCWCF-UHFFFAOYSA-N | $5.8\times10^{-2}$ | | Burkhard et al. (1985) | V | |
| | $4.5\times10^{-2}$ | | Keshavarz et al. (2022) | Q | |
| | $7.7\times10^{-2}$ | | Duchowicz et al. (2020) | Q | 184 |
| | $2.0\times10^{-1}$ | | Hilal et al. (2008) | Q | |
| | $1.3\times10^{-1}$ | | Modarresi et al. (2007) | Q | 67 |
| | $3.9\times10^{-2}$ | | Fang Lee (2007) | Q | 721 |
| | $6.5\times10^{-2}$ | | Fang Lee (2007) | Q | 722 |
| | $4.6\times10^{-2}$ | | Dunnivant et al. (1992) | Q | |
| | $4.7\times10^{-2}$ | | Sabljić and Güsten (1989) | Q | |
| | $9.9\times10^{-2}$ | | Duchowicz et al. (2020) | ? | 185, 21 |
| 2,4,4',6-tetrachlorobiphenyl | $2.1\times10^{-2}$ | 6400 | Paasivirta and Sinkkonen (2009) | V | |
| $C_{12}H_6Cl_4$ | $2.4\times10^{-2}$ | | Burkhard et al. (1985) | V | |
| (PCB-75) | $2.4\times10^{-2}$ | | Fang Lee (2007) | Q | 721 |
| [32598-12-2] | $4.4\times10^{-2}$ | | Fang Lee (2007) | Q | 722 |
| RZFZBHKDGHISSH-UHFFFAOYSA-N | $2.1\times10^{-2}$ | | Dunnivant et al. (1992) | Q | |
| | $1.8\times10^{-2}$ | | Sabljić and Güsten (1989) | Q | |
| 2,3',4',5'-tetrachlorobiphenyl | $7.7\times10^{-2}$ | | Murphy et al. (1987) | M | 12 |
| $C_{12}H_6Cl_4$ | $2.3\times10^{-3}$ | 5500 | Paasivirta and Sinkkonen (2009) | V | |
| (PCB-76) | $8.2\times10^{-2}$ | | Burkhard et al. (1985) | V | |
| [70362-48-0] | $1.4\times10^{-1}$ | | Fang Lee (2007) | Q | 721 |
| QILUYCYPNYWMIL-UHFFFAOYSA-N | $7.0\times10^{-2}$ | | Fang Lee (2007) | Q | 722 |
| | $4.1\times10^{-2}$ | | Dunnivant et al. (1992) | Q | |
| | $4.4\times10^{-2}$ | | Sabljić and Güsten (1989) | Q | |
| 3,3',4,4'-tetrachlorobiphenyl | $1.2\times10^{-2}$ | | Bhangare et al. (2019) | M | 725 |
| $C_{12}H_6Cl_4$ | $7.1\times10^{-2}$ | | Bhangare et al. (2019) | M | 726 |
| (PCB-77) | $3.1\times10^{-2}$ | | Lau et al. (2006) | M | 719 |
| [32598-13-3] | $1.8\times10^{-2}$ | | Lau et al. (2006) | M | 720 |
| UQMGJOKDKOLIDP-UHFFFAOYSA-N | $9.1\times10^{-2}$ | | Fang et al. (2006) | M | |
| | $2.9\times10^{-2}$ | 13000 | Charles and Destaillats (2005) | M | 33 |
| | $6.2\times10^{-2}$ | 4800 | Bamford et al. (2000) | M | |
| | $1.0\times10^{-1}$ | | Dunnivant et al. (1988) | M | |
| | $1.0\times10^{-1}$ | | Dunnivant and Elzerman (1988) | M | 723 |
| | $6.0\times10^{-4}$ | 4600 | Paasivirta and Sinkkonen (2009) | V | |
| | $5.8\times10^{-2}$ | | Mackay et al. (2006b) | V | |
| | $5.8\times10^{-1}$ | | Mackay et al. (1992a) | V | |



Table A6.3: Polychlorinated biphenyls (PCBs) (...continued)

| Substance Formula (Trivial Name) [CAS Registry Number] InChIKey | $H_s^{cp}$ (at $T^\ominus$) $\left[\dfrac{\mathrm{mol}}{\mathrm{m^3\,Pa}}\right]$ | $\dfrac{\mathrm{d}\ln H_s^{cp}}{\mathrm{d}(1/T)}$ [K] | Reference | Type | Note |
|---|---|---|---|---|---|
| | $5.9\times10^{-1}$ | | Shiu and Mackay (1986) | V | |
| | $2.3\times10^{-1}$ | | Burkhard et al. (1985) | V | |
| | $8.3\times10^{-3}$ | 7400 | Paasivirta et al. (1999) | T | |
| | $2.4\times10^{-1}$ | | Keshavarz et al. (2022) | Q | |
| | $1.9\times10^{-1}$ | | Duchowicz et al. (2020) | Q | |
| | $3.6\times10^{-1}$ | | Hilal et al. (2008) | Q | |
| | $1.2\times10^{-1}$ | | Modarresi et al. (2007) | Q | 67 |
| | $9.4\times10^{-2}$ | | Fang Lee (2007) | Q | 721 |
| | $8.0\times10^{-2}$ | | Fang Lee (2007) | Q | 722 |
| | | 6100 | Kühne et al. (2005) | Q | |
| | $9.6\times10^{-2}$ | | Dunnivant et al. (1992) | Q | |
| | $7.9\times10^{-2}$ | | Meylan and Howard (1991) | Q | |
| | 1.0 | | Duchowicz et al. (2020) | ? | 185, 21 |
| | | 5600 | Kühne et al. (2005) | ? | |
| 3,3',4,5-tetrachlorobiphenyl $C_{12}H_6Cl_4$ (PCB-78) [70362-49-1] SXFLURRQRFKBNN-UHFFFAOYSA-N | $5.1\times10^{-3}$ | 5600 | Paasivirta and Sinkkonen (2009) | V | |
| | $1.7\times10^{-1}$ | | Burkhard et al. (1985) | V | |
| | $1.7\times10^{-1}$ | | Fang Lee (2007) | Q | 721 |
| | $7.5\times10^{-2}$ | | Fang Lee (2007) | Q | 722 |
| | $6.0\times10^{-2}$ | | Dunnivant et al. (1992) | Q | |
| | $4.4\times10^{-2}$ | | Sabljić and Güsten (1989) | Q | |
| 3,3',4,5'-tetrachlorobiphenyl $C_{12}H_6Cl_4$ (PCB-79) [41464-48-6] QLCTXEMDCZGPCG-UHFFFAOYSA-N | $3.8\times10^{-3}$ | 5400 | Paasivirta and Sinkkonen (2009) | V | |
| | $1.4\times10^{-1}$ | | Burkhard et al. (1985) | V | |
| | $2.4\times10^{-1}$ | | Keshavarz et al. (2022) | Q | |
| | $1.1\times10^{-1}$ | | Duchowicz et al. (2020) | Q | 299 |
| | $2.5\times10^{-1}$ | | Hilal et al. (2008) | Q | |
| | $8.2\times10^{-2}$ | | Modarresi et al. (2007) | Q | 67 |
| | $1.9\times10^{-1}$ | | Fang Lee (2007) | Q | 721 |
| | $6.3\times10^{-2}$ | | Fang Lee (2007) | Q | 722 |
| | $5.0\times10^{-2}$ | | Dunnivant et al. (1992) | Q | |
| | $2.9\times10^{-2}$ | | Sabljić and Güsten (1989) | Q | |
| | $1.1\times10^{-1}$ | | Duchowicz et al. (2020) | ? | 185, 21 |
| 3,3',5,5'-tetrachlorobiphenyl $C_{12}H_6Cl_4$ (PCB-80) [33284-52-5] UTMWFJSRHLYRPY-UHFFFAOYSA-N | $9.4\times10^{-4}$ | 5100 | Paasivirta and Sinkkonen (2009) | V | |
| | $8.0\times10^{-2}$ | | Burkhard et al. (1985) | V | |
| | $2.6\times10^{-1}$ | | Fang Lee (2007) | Q | 721 |
| | $5.2\times10^{-2}$ | | Fang Lee (2007) | Q | 722 |
| | $2.6\times10^{-2}$ | | Dunnivant et al. (1992) | Q | |
| | $1.6\times10^{-2}$ | | Sabljić and Güsten (1989) | Q | |
| 3,4,4',5-tetrachlorobiphenyl $C_{12}H_6Cl_4$ (PCB-81) [70362-50-4] BHWVLZJTVIYLIV-UHFFFAOYSA-N | $8.8\times10^{-2}$ | | Fang et al. (2006) | M | |
| | $4.1\times10^{-2}$ | 4000 | Bamford et al. (2002) | M | |
| | $2.0\times10^{-3}$ | 5300 | Paasivirta and Sinkkonen (2009) | V | |
| | $1.9\times10^{-1}$ | | Burkhard et al. (1985) | V | |
| | $8.6\times10^{-2}$ | | Fang Lee (2007) | Q | 721 |
| | $7.2\times10^{-2}$ | | Fang Lee (2007) | Q | 722 |
| | $6.9\times10^{-2}$ | | Dunnivant et al. (1992) | Q | |
| | $6.7\times10^{-2}$ | | Sabljić and Güsten (1989) | Q | |



Table A6.3: Polychlorinated biphenyls (PCBs) (. . . continued)

| Substance Formula (Trivial Name) [CAS Registry Number] InChIKey | $H_s^{cp}$ (at $T^{\ominus}$) $\left[\dfrac{\text{mol}}{\text{m}^3\,\text{Pa}}\right]$ | $\dfrac{\text{d}\ln H_s^{cp}}{\text{d}(1/T)}$ [K] | Reference | Type | Note |
|---|---|---|---|---|---|
| 2,2',3,3',4-pentachlorobiphenyl | $2.7 \times 10^{-2}$ | 5100 | Bamford et al. (2002) | M | |
| $C_{12}H_5Cl_5$ | $8.4 \times 10^{-2}$ | | Murphy et al. (1987) | M | 12 |
| (PCB-82) | $4.9 \times 10^{-2}$ | | Murphy et al. (1983a) | M | 24 |
| [52663-62-4] | $3.2 \times 10^{-3}$ | 5800 | Paasivirta and Sinkkonen (2009) | V | |
| AUGNBQPSMWGAJE-UHFFFAOYSA-N | $5.0 \times 10^{-2}$ | | Shiu and Mackay (1986) | V | |
| | $8.0 \times 10^{-2}$ | | Burkhard et al. (1985) | V | |
| | $1.6 \times 10^{-1}$ | | Fang Lee (2007) | Q | 721 |
| | $1.5 \times 10^{-1}$ | | Fang Lee (2007) | Q | 722 |
| | $6.7 \times 10^{-2}$ | | Dunnivant et al. (1992) | Q | |
| | $8.1 \times 10^{-2}$ | | Sabljić and Güsten (1989) | Q | |
| 2,2',3,3',5-pentachlorobiphenyl | $2.3 \times 10^{-2}$ | 3600 | Bamford et al. (2002) | M | |
| $C_{12}H_5Cl_5$ | $6.0 \times 10^{-2}$ | | Murphy et al. (1987) | M | 12 |
| (PCB-83) | $7.7 \times 10^{-3}$ | 6300 | Paasivirta and Sinkkonen (2009) | V | |
| [60145-20-2] | $4.7 \times 10^{-2}$ | | Burkhard et al. (1985) | V | |
| SUBRHHYLRGOTHL-UHFFFAOYSA-N | $2.2 \times 10^{-1}$ | | Fang Lee (2007) | Q | 721 |
| | $1.4 \times 10^{-1}$ | | Fang Lee (2007) | Q | 722 |
| | $4.7 \times 10^{-2}$ | | Dunnivant et al. (1992) | Q | |
| | $3.8 \times 10^{-2}$ | | Sabljić and Güsten (1989) | Q | |
| 2,2',3,3',6-pentachlorobiphenyl | $5.7 \times 10^{-2}$ | | Murphy et al. (1987) | M | 12 |
| $C_{12}H_5Cl_5$ | $2.3 \times 10^{-3}$ | 6000 | Paasivirta and Sinkkonen (2009) | V | |
| (PCB-84) | $1.9 \times 10^{-2}$ | | Burkhard et al. (1985) | V | |
| [52663-60-2] | $1.3 \times 10^{-1}$ | | Fang Lee (2007) | Q | 721 |
| QVWUJLANSDKRAH-UHFFFAOYSA-N | $1.2 \times 10^{-1}$ | | Fang Lee (2007) | Q | 722 |
| | $3.9 \times 10^{-2}$ | | Dunnivant et al. (1992) | Q | |
| | $4.3 \times 10^{-2}$ | | Sabljić and Güsten (1989) | Q | |
| 2,2',3,4,4'-pentachlorobiphenyl | $2.3 \times 10^{-2}$ | 3100 | Bamford et al. (2002) | M | |
| $C_{12}H_5Cl_5$ | $1.5 \times 10^{-1}$ | | Brunner et al. (1990) | M | |
| (PCB-85) | $6.0 \times 10^{-2}$ | | Murphy et al. (1987) | M | 12 |
| [65510-45-4] | $2.8 \times 10^{-2}$ | 6600 | Paasivirta and Sinkkonen (2009) | V | |
| LACXVZHAJMVESG-UHFFFAOYSA-N | $5.6 \times 10^{-2}$ | | Burkhard et al. (1985) | V | |
| | $2.4 \times 10^{-1}$ | | Keshavarz et al. (2022) | Q | |
| | $1.6 \times 10^{-1}$ | | Duchowicz et al. (2020) | Q | |
| | $9.2 \times 10^{-2}$ | | Hilal et al. (2008) | Q | |
| | $1.2 \times 10^{-1}$ | | Modarresi et al. (2007) | Q | 67 |
| | $7.8 \times 10^{-2}$ | | Fang Lee (2007) | Q | 721 |
| | $1.1 \times 10^{-1}$ | | Fang Lee (2007) | Q | 722 |
| | $5.1 \times 10^{-2}$ | | Dunnivant et al. (1992) | Q | |
| | $4.0 \times 10^{-2}$ | | Sabljić and Güsten (1989) | Q | |
| | $1.5 \times 10^{-1}$ | | Duchowicz et al. (2020) | ? | 185, 21 |



Table A6.3: Polychlorinated biphenyls (PCBs) (...continued)

| Substance Formula (Trivial Name) [CAS Registry Number] InChIKey | $H_s^{cp}$ (at $T^\ominus$) $\left[\dfrac{\mathrm{mol}}{\mathrm{m}^3\,\mathrm{Pa}}\right]$ | $\dfrac{\mathrm{d}\ln H_s^{cp}}{\mathrm{d}(1/T)}$ [K] | Reference | Type | Note |
|---|---|---|---|---|---|
| 2,2',3,4,5-pentachlorobiphenyl | $3.2\times10^{-3}$ | | Duchowicz et al. (2020) | V | 186 |
| $C_{12}H_5Cl_5$ | $8.9\times10^{-3}$ | 6500 | Paasivirta and Sinkkonen (2009) | V | |
| (PCB-86) | $6.6\times10^{-3}$ | | Mackay et al. (2006b) | V | |
| [55312-69-1] | $6.6\times10^{-3}$ | | Mackay et al. (1992a) | V | |
| AIURIRUDHVDRFQ-UHFFFAOYSA-N | $6.6\times10^{-3}$ | | Shiu and Mackay (1986) | V | |
| | $1.2\times10^{-2}$ | | Burkhard et al. (1985) | V | |
| | $1.4\times10^{-1}$ | | Duchowicz et al. (2020) | Q | |
| | $7.5\times10^{-2}$ | | Hilal et al. (2008) | Q | |
| | $1.4\times10^{-1}$ | | Fang Lee (2007) | Q | 721 |
| | $1.2\times10^{-1}$ | | Fang Lee (2007) | Q | 722 |
| | $4.1\times10^{-2}$ | | Dunnivant et al. (1992) | Q | |
| | $5.8\times10^{-2}$ | | Sabljić and Güsten (1989) | Q | |
| 2,2',3,4,5'-pentachlorobiphenyl | $2.7\times10^{-2}$ | 3900 | Bamford et al. (2000) | M | |
| $C_{12}H_5Cl_5$ | $7.8\times10^{-2}$ | | Murphy et al. (1987) | M | 12 |
| (PCB-87) | $1.4\times10^{-1}$ | | Brownawell (1986) | M | 294 |
| [38380-02-8] | $3.0\times10^{-2}$ | | Murphy et al. (1983a) | M | 24 |
| OPKYDBFRKPQCBS-UHFFFAOYSA-N | $4.1\times10^{-3}$ | 6000 | Paasivirta and Sinkkonen (2009) | V | |
| | $4.0\times10^{-2}$ | | Mackay et al. (2006b) | V | |
| | $4.0\times10^{-2}$ | | Mackay et al. (1992a) | V | |
| | $4.0\times10^{-2}$ | | Shiu and Mackay (1986) | V | |
| | $5.0\times10^{-2}$ | | Burkhard et al. (1985) | V | |
| | $2.4\times10^{-1}$ | | Keshavarz et al. (2022) | Q | |
| | $1.5\times10^{-1}$ | | Duchowicz et al. (2020) | Q | 299 |
| | $2.3\times10^{-2}$ | | Bhangare et al. (2019) | Q | |
| | $1.2\times10^{-1}$ | | Hilal et al. (2008) | Q | |
| | $1.6\times10^{-1}$ | | Fang Lee (2007) | Q | 721 |
| | $1.0\times10^{-1}$ | | Fang Lee (2007) | Q | 722 |
| | | 5000 | Kühne et al. (2005) | Q | |
| | $5.4\times10^{-2}$ | | Dunnivant et al. (1992) | Q | |
| | $5.5\times10^{-2}$ | | Sabljić and Güsten (1989) | Q | |
| | $1.3\times10^{-1}$ | | Duchowicz et al. (2020) | ? | 185, 21 |
| | | 4200 | Kühne et al. (2005) | ? | |
| 2,2',3,4,6-pentachlorobiphenyl | $9.6\times10^{-3}$ | 6800 | Paasivirta and Sinkkonen (2009) | V | |
| $C_{12}H_5Cl_5$ | $7.3\times10^{-3}$ | | Burkhard et al. (1985) | V | |
| (PCB-88) | $1.2\times10^{-1}$ | | Fang Lee (2007) | Q | 721 |
| [55215-17-3] | $7.8\times10^{-2}$ | | Fang Lee (2007) | Q | 722 |
| QGDKRLQRLFUJPP-UHFFFAOYSA-N | $2.6\times10^{-2}$ | | Dunnivant et al. (1992) | Q | |
| | $2.9\times10^{-2}$ | | Sabljić and Güsten (1989) | Q | |
| 2,2',3,4,6'-pentachlorobiphenyl | $2.2\times10^{-2}$ | 2500 | Bamford et al. (2002) | M | |
| $C_{12}H_5Cl_5$ | $4.3\times10^{-3}$ | 6100 | Paasivirta and Sinkkonen (2009) | V | |
| (PCB-89) | $2.4\times10^{-2}$ | | Burkhard et al. (1985) | V | |
| [73575-57-2] | $8.7\times10^{-2}$ | | Fang Lee (2007) | Q | 721 |
| GLOOIONSKMZYQZ-UHFFFAOYSA-N | $9.8\times10^{-2}$ | | Fang Lee (2007) | Q | 722 |
| | $3.3\times10^{-2}$ | | Dunnivant et al. (1992) | Q | |
| | $3.4\times10^{-2}$ | | Sabljić and Güsten (1989) | Q | |



Table A6.3: Polychlorinated biphenyls (PCBs) (. . . continued)

| Substance Formula (Trivial Name) [CAS Registry Number] InChIKey | $H_s^{cp}$ (at $T^\ominus$) $\left[\dfrac{\mathrm{mol}}{\mathrm{m}^3\,\mathrm{Pa}}\right]$ | $\dfrac{\mathrm{d}\ln H_s^{cp}}{\mathrm{d}(1/T)}$ [K] | Reference | Type | Note |
|---|---|---|---|---|---|
| 2,2',3,4',5-pentachlorobiphenyl | $2.1\times10^{-2}$ | 6600 | Paasivirta and Sinkkonen (2009) | V | |
| $C_{12}H_5Cl_5$ | $3.3\times10^{-2}$ | | Burkhard et al. (1985) | V | |
| (PCB-90) | $1.1\times10^{-1}$ | | Fang Lee (2007) | Q | 721 |
| [68194-07-0] | $8.8\times10^{-2}$ | | Fang Lee (2007) | Q | 722 |
| SUOAMBOBSWRMNQ-UHFFFAOYSA-N | $3.4\times10^{-2}$ | | Dunnivant et al. (1992) | Q | |
| | $2.6\times10^{-2}$ | | Sabljić and Güsten (1989) | Q | |
| 2,2',3,4',6-pentachlorobiphenyl | $1.9\times10^{-2}$ | 1200 | Bamford et al. (2002) | M | |
| $C_{12}H_5Cl_5$ | $3.6\times10^{-2}$ | | Murphy et al. (1987) | M | 12 |
| (PCB-91) | $1.1\times10^{-2}$ | 6500 | Paasivirta and Sinkkonen (2009) | V | |
| [68194-05-8] | $1.4\times10^{-2}$ | | Burkhard et al. (1985) | V | |
| CXKIGWXPPVZSQK-UHFFFAOYSA-N | $6.6\times10^{-2}$ | | Fang Lee (2007) | Q | 721 |
| | $8.3\times10^{-2}$ | | Fang Lee (2007) | Q | 722 |
| | $2.9\times10^{-2}$ | | Dunnivant et al. (1992) | Q | |
| | $2.8\times10^{-2}$ | | Sabljić and Güsten (1989) | Q | |
| 2,2',3,5,5'-pentachlorobiphenyl | $2.2\times10^{-2}$ | 2900 | Bamford et al. (2002) | M | |
| $C_{12}H_5Cl_5$ | $1.2\times10^{-2}$ | 6500 | Paasivirta and Sinkkonen (2009) | V | |
| (PCB-92) | $3.0\times10^{-2}$ | | Burkhard et al. (1985) | V | |
| [52663-61-3] | $2.2\times10^{-1}$ | | Fang Lee (2007) | Q | 721 |
| CRCBRZBVCDKPGA-UHFFFAOYSA-N | $9.5\times10^{-2}$ | | Fang Lee (2007) | Q | 722 |
| | $3.8\times10^{-2}$ | | Dunnivant et al. (1992) | Q | |
| | $3.1\times10^{-2}$ | | Sabljić and Güsten (1989) | Q | |
| 2,2',3,5,6-pentachlorobiphenyl | $4.1\times10^{-3}$ | 6500 | Paasivirta and Sinkkonen (2009) | V | |
| $C_{12}H_5Cl_5$ | $8.3\times10^{-3}$ | | Burkhard et al. (1985) | V | |
| (PCB-93) | $1.2\times10^{-1}$ | | Fang Lee (2007) | Q | 721 |
| [73575-56-1] | $1.3\times10^{-1}$ | | Fang Lee (2007) | Q | 722 |
| BMXRLHMJGHJGLR-UHFFFAOYSA-N | $2.9\times10^{-2}$ | | Dunnivant et al. (1992) | Q | |
| | $3.5\times10^{-2}$ | | Sabljić and Güsten (1989) | Q | |
| 2,2',3,5,6'-pentachlorobiphenyl | $4.5\times10^{-3}$ | 6300 | Paasivirta and Sinkkonen (2009) | V | |
| $C_{12}H_5Cl_5$ | $1.4\times10^{-2}$ | | Burkhard et al. (1985) | V | |
| (PCB-94) | $1.2\times10^{-1}$ | | Fang Lee (2007) | Q | 721 |
| [73575-55-0] | $9.1\times10^{-2}$ | | Fang Lee (2007) | Q | 722 |
| FJUVPPYNSDTRQV-UHFFFAOYSA-N | $2.5\times10^{-2}$ | | Dunnivant et al. (1992) | Q | |
| | $2.3\times10^{-2}$ | | Sabljić and Güsten (1989) | Q | |
| 2,2',3,5',6-pentachlorobiphenyl | $2.1\times10^{-2}$ | 2500 | Bamford et al. (2002) | M | |
| $C_{12}H_5Cl_5$ | $5.0\times10^{-2}$ | | Murphy et al. (1987) | M | 12 |
| (PCB-95) | $3.3\times10^{-3}$ | 6200 | Paasivirta and Sinkkonen (2009) | V | |
| [38379-99-6] | $1.2\times10^{-2}$ | | Burkhard et al. (1985) | V | |
| GXNNLIMMEXHBKV-UHFFFAOYSA-N | $2.4\times10^{-1}$ | | Keshavarz et al. (2022) | Q | |
| | $7.9\times10^{-2}$ | | Duchowicz et al. (2020) | Q | |
| | $1.3\times10^{-1}$ | | Fang Lee (2007) | Q | 721 |
| | $9.0\times10^{-2}$ | | Fang Lee (2007) | Q | 722 |
| | $3.3\times10^{-2}$ | | Dunnivant et al. (1992) | Q | |
| | $3.4\times10^{-2}$ | | Sabljić and Güsten (1989) | Q | |
| | $8.2\times10^{-2}$ | | Duchowicz et al. (2020) | ? | 185, 21 |



Table A6.3: Polychlorinated biphenyls (PCBs) (...continued)

| Substance Formula (Trivial Name) [CAS Registry Number] InChIKey | $H_s^{cp}$ (at $T^\ominus$) $\left[\dfrac{\text{mol}}{\text{m}^3\,\text{Pa}}\right]$ | $\dfrac{\text{d}\ln H_s^{cp}}{\text{d}(1/T)}$ [K] | Reference | Type | Note |
|---|---|---|---|---|---|
| 2,2',3,6,6'-pentachlorobiphenyl | $8.7\times10^{-4}$ | 5800 | Paasivirta and Sinkkonen (2009) | V | |
| $C_{12}H_5Cl_5$ | $7.2\times10^{-3}$ | | Burkhard et al. (1985) | V | |
| (PCB-96) | $7.4\times10^{-2}$ | | Fang Lee (2007) | Q | 721 |
| [73575-54-9] | $6.5\times10^{-2}$ | | Fang Lee (2007) | Q | 722 |
| QQFGAXUIQVKBKU-UHFFFAOYSA-N | $2.4\times10^{-2}$ | | Dunnivant et al. (1992) | Q | |
| | $2.6\times10^{-2}$ | | Sabljić and Güsten (1989) | Q | |
| 2,2',3,4',5'-pentachlorobiphenyl | $2.3\times10^{-2}$ | 3600 | Bamford et al. (2002) | M | |
| $C_{12}H_5Cl_5$ | $1.3\times10^{-1}$ | | Brunner et al. (1990) | M | |
| (PCB-97) | $6.6\times10^{-2}$ | | Murphy et al. (1987) | M | 12 |
| [41464-51-1] | $8.6\times10^{-3}$ | 6300 | Paasivirta and Sinkkonen (2009) | V | |
| JTUSORDQZVOEAZ-UHFFFAOYSA-N | $4.8\times10^{-2}$ | | Burkhard et al. (1985) | V | |
| | $2.4\times10^{-1}$ | | Keshavarz et al. (2022) | Q | |
| | $1.5\times10^{-1}$ | | Duchowicz et al. (2020) | Q | |
| | $1.5\times10^{-1}$ | | Hilal et al. (2008) | Q | |
| | $1.4\times10^{-1}$ | | Modarresi et al. (2007) | Q | 67 |
| | $1.2\times10^{-1}$ | | Fang Lee (2007) | Q | 721 |
| | $1.1\times10^{-1}$ | | Fang Lee (2007) | Q | 722 |
| | $5.5\times10^{-2}$ | | Dunnivant et al. (1992) | Q | |
| | $5.5\times10^{-2}$ | | Sabljić and Güsten (1989) | Q | |
| | $1.3\times10^{-1}$ | | Duchowicz et al. (2020) | ? | 185, 21 |
| 2,2',3,4',6-pentachlorobiphenyl | $5.5\times10^{-3}$ | 6300 | Paasivirta and Sinkkonen (2009) | V | |
| $C_{12}H_5Cl_5$ | $1.5\times10^{-2}$ | | Burkhard et al. (1985) | V | |
| (PCB-98) | $7.6\times10^{-2}$ | | Fang Lee (2007) | Q | 721 |
| [60233-25-2] | $6.5\times10^{-2}$ | | Fang Lee (2007) | Q | 722 |
| GOFFZTAPOOICFT-UHFFFAOYSA-N | $2.5\times10^{-2}$ | | Dunnivant et al. (1992) | Q | |
| | $2.0\times10^{-2}$ | | Sabljić and Güsten (1989) | Q | |
| 2,2',4,4',5-pentachlorobiphenyl | $2.2\times10^{-2}$ | | Lau et al. (2006) | M | 719 |
| $C_{12}H_5Cl_5$ | $4.2\times10^{-3}$ | | Lau et al. (2006) | M | 720 |
| (PCB-99) | $8.8\times10^{-3}$ | 8700 | Charles and Destaillats (2005) | M | 33 |
| [38380-01-7] | $2.1\times10^{-2}$ | 1900 | Bamford et al. (2002) | M | |
| LMQJBFRGXHMNOX-UHFFFAOYSA-N | $1.3\times10^{-1}$ | | Brunner et al. (1990) | M | |
| | $4.6\times10^{-2}$ | | Murphy et al. (1987) | M | 12 |
| | $2.1\times10^{-2}$ | 6600 | Paasivirta and Sinkkonen (2009) | V | |
| | $3.4\times10^{-2}$ | | Burkhard et al. (1985) | V | |
| | $2.4\times10^{-1}$ | | Keshavarz et al. (2022) | Q | |
| | $8.6\times10^{-2}$ | | Duchowicz et al. (2020) | Q | 299 |
| | $6.2\times10^{-2}$ | | Fang Lee (2007) | Q | 721 |
| | $7.9\times10^{-2}$ | | Fang Lee (2007) | Q | 722 |
| | $4.0\times10^{-2}$ | | Dunnivant et al. (1992) | Q | |
| | $3.3\times10^{-2}$ | | Sabljić and Güsten (1989) | Q | |
| | $1.3\times10^{-1}$ | | Duchowicz et al. (2020) | ? | 185, 21 |





Table A6.3: Polychlorinated biphenyls (PCBs) (. . . continued)

| Substance<br>Formula<br>(Trivial Name)<br>[CAS Registry Number]<br>InChIKey | $H_s^{cp}$<br>(at $T^{\ominus}$)<br>$\left[\dfrac{\mathrm{mol}}{\mathrm{m^3\,Pa}}\right]$ | $\dfrac{\mathrm{d}\ln H_s^{cp}}{\mathrm{d}(1/T)}$<br><br>[K] | Reference | Type | Note |
|---|---|---|---|---|---|
| 2,2',4,4',6-pentachlorobiphenyl | $9.7\times10^{-3}$ | 6600 | Paasivirta and Sinkkonen (2009) | V | |
| $C_{12}H_5Cl_5$ | $1.0\times10^{-2}$ | | Burkhard et al. (1985) | V | |
| (PCB-100) | $3.8\times10^{-2}$ | | Fang Lee (2007) | Q | 721 |
| [39485-83-1] | $4.6\times10^{-2}$ | | Fang Lee (2007) | Q | 722 |
| RKUAZJIXKHPFRK-UHFFFAOYSA-N | $1.8\times10^{-2}$ | | Dunnivant et al. (1992) | Q | |
| | $1.6\times10^{-2}$ | | Sabljić and Güsten (1989) | Q | |
| 2,2',4,5,5'-pentachlorobiphenyl | $3.2\times10^{-2}$ | 6800 | Li et al. (2003) | L | 366 |
| $C_{12}H_5Cl_5$ | $4.1\times10^{-2}$ | 7500 | Li et al. (2003) | L | 367 |
| (PCB-101) | $6.9\times10^{-3}$ | | Bhangare et al. (2019) | M | 725 |
| [37680-73-2] | $2.0\times10^{-2}$ | | Bhangare et al. (2019) | M | 726 |
| LAHWLEDBADHJGA-UHFFFAOYSA-N | $2.4\times10^{-2}$ | 3600 | Bamford et al. (2000) | M | |
| | $3.9\times10^{-2}$ | | Dunnivant et al. (1988) | M | |
| | $3.9\times10^{-2}$ | | Dunnivant and Elzerman (1988) | M | 723 |
| | $5.5\times10^{-2}$ | | Murphy et al. (1987) | M | 12 |
| | $1.4\times10^{-1}$ | | Oliver (1985) | M | |
| | | | Westcott et al. (1981) | M | 728 |
| | $8.9\times10^{-3}$ | 6400 | Paasivirta and Sinkkonen (2009) | V | |
| | $2.8\times10^{-2}$ | | Mackay et al. (2006b) | V | |
| | $2.8\times10^{-2}$ | | Mackay et al. (1992a) | V | |
| | $2.9\times10^{-2}$ | | Shiu and Mackay (1986) | V | |
| | $3.1\times10^{-2}$ | | Burkhard et al. (1985) | V | |
| | $2.0\times10^{-2}$ | 8100 | Paasivirta et al. (1999) | T | |
| | $2.4\times10^{-1}$ | | Keshavarz et al. (2022) | Q | |
| | $7.9\times10^{-2}$ | | Duchowicz et al. (2020) | Q | 184 |
| | $1.2\times10^{-1}$ | | Hilal et al. (2008) | Q | |
| | $1.2\times10^{-1}$ | | Modarresi et al. (2007) | Q | 67 |
| | $1.3\times10^{-1}$ | | Fang Lee (2007) | Q | 721 |
| | $7.9\times10^{-2}$ | | Fang Lee (2007) | Q | 722 |
| | | 4600 | Kühne et al. (2005) | Q | |
| | $1.2\times10^{-1}$ | | Yaffe et al. (2003) | Q | 248, 249 |
| | $6.1\times10^{-2}$ | | English and Carroll (2001) | Q | 230, 231 |
| | $4.0\times10^{-2}$ | | Dunnivant et al. (1992) | Q | |
| | $1.1\times10^{-1}$ | | Meylan and Howard (1991) | Q | |
| | $1.1\times10^{-1}$ | | Duchowicz et al. (2020) | ? | 185, 21 |
| | | 3900 | Kühne et al. (2005) | ? | |
| 2,2',4,5,6'-pentachlorobiphenyl | $1.1\times10^{-1}$ | | Brunner et al. (1990) | M | |
| $C_{12}H_5Cl_5$ | $6.3\times10^{-3}$ | 6300 | Paasivirta and Sinkkonen (2009) | V | |
| (PCB-102) | $1.5\times10^{-2}$ | | Burkhard et al. (1985) | V | |
| [68194-06-9] | $2.4\times10^{-1}$ | | Keshavarz et al. (2022) | Q | |
| BWWVXHRLMPBDCK-UHFFFAOYSA-N | $8.6\times10^{-2}$ | | Duchowicz et al. (2020) | Q | |
| | $8.8\times10^{-2}$ | | Hilal et al. (2008) | Q | |
| | $1.3\times10^{-1}$ | | Modarresi et al. (2007) | Q | 67 |
| | $6.9\times10^{-2}$ | | Fang Lee (2007) | Q | 721 |
| | $7.7\times10^{-2}$ | | Fang Lee (2007) | Q | 722 |
| | $2.7\times10^{-2}$ | | Dunnivant et al. (1992) | Q | |
| | $2.8\times10^{-2}$ | | Sabljić and Güsten (1989) | Q | |





Table A6.3: Polychlorinated biphenyls (PCBs) (...continued)

| Substance Formula (Trivial Name) [CAS Registry Number] InChIKey | $H_s^{cp}$ (at $T^\ominus$) $\left[\dfrac{\mathrm{mol}}{\mathrm{m^3\,Pa}}\right]$ | $\dfrac{\mathrm{d}\ln H_s^{cp}}{\mathrm{d}(1/T)}$ [K] | Reference | Type | Note |
|---|---|---|---|---|---|
| | $1.1\times10^{-1}$ | | Duchowicz et al. (2020) | ? | 185, 21 |
| 2,2',4,5',6-pentachlorobiphenyl | $8.1\times10^{-3}$ | 6600 | Paasivirta and Sinkkonen (2009) | V | |
| $C_{12}H_5Cl_5$ | $9.1\times10^{-3}$ | | Burkhard et al. (1985) | V | |
| (PCB-103) | $7.7\times10^{-2}$ | | Fang Lee (2007) | Q | 721 |
| [60145-21-3] | $4.6\times10^{-2}$ | | Fang Lee (2007) | Q | 722 |
| PQHZWWBJPCNNGI-UHFFFAOYSA-N | $2.0\times10^{-2}$ | | Dunnivant et al. (1992) | Q | |
| | $1.8\times10^{-2}$ | | Sabljić and Güsten (1989) | Q | |
| 2,2',4,6,6'-pentachlorobiphenyl | $1.5\times10^{-2}$ | 1700 | Bamford et al. (2000) | M | |
| $C_{12}H_5Cl_5$ | $1.1\times10^{-2}$ | | Dunnivant et al. (1988) | M | |
| (PCB-104) | $1.1\times10^{-2}$ | | Dunnivant and Elzerman (1988) | M | 723 |
| [56558-16-8] | $2.8\times10^{-3}$ | 6300 | Paasivirta and Sinkkonen (2009) | V | |
| MTCPZNVSDFCBBE-UHFFFAOYSA-N | $4.3\times10^{-2}$ | | Mackay et al. (2006b) | V | |
| | $7.2\times10^{-2}$ | | Mackay et al. (1992a) | V | |
| | $5.4\times10^{-3}$ | | Burkhard et al. (1985) | V | |
| | $2.0\times10^{-2}$ | | Bhangare et al. (2019) | Q | |
| | $4.2\times10^{-2}$ | | Fang Lee (2007) | Q | 721 |
| | $3.4\times10^{-2}$ | | Fang Lee (2007) | Q | 722 |
| | | 3100 | Kühne et al. (2005) | Q | |
| | $1.3\times10^{-2}$ | | Dunnivant et al. (1992) | Q | |
| | | 2000 | Kühne et al. (2005) | ? | |
| 2,3,3',4,4'-pentachlorobiphenyl | $3.0\times10^{-2}$ | 6800 | Li et al. (2003) | L | 366 |
| $C_{12}H_5Cl_5$ | $7.2\times10^{-2}$ | 7500 | Li et al. (2003) | L | 367 |
| (PCB-105) | $1.8\times10^{-1}$ | | Fang et al. (2006) | M | |
| [32598-14-4] | $3.0\times10^{-2}$ | 9100 | Bamford et al. (2000) | M | |
| WIDHRBRBACOVOY-UHFFFAOYSA-N | $1.2\times10^{-2}$ | | Duchowicz et al. (2020) | V | 186 |
| | $5.0\times10^{-3}$ | 5700 | Paasivirta and Sinkkonen (2009) | V | |
| | $1.8\times10^{-1}$ | | Burkhard et al. (1985) | V | |
| | $2.9\times10^{-2}$ | 8300 | Paasivirta et al. (1999) | T | |
| | $2.7\times10^{-1}$ | | Duchowicz et al. (2020) | Q | |
| | $2.0\times10^{-2}$ | | Bhangare et al. (2019) | Q | |
| | $9.7\times10^{-2}$ | | Fang Lee (2007) | Q | 721 |
| | $1.4\times10^{-1}$ | | Fang Lee (2007) | Q | 722 |
| | $9.9\times10^{-2}$ | | Dunnivant et al. (1992) | Q | |
| | $1.6\times10^{-1}$ | | Sabljić and Güsten (1989) | Q | |
| 2,3,3',4,5-pentachlorobiphenyl | $1.3\times10^{-2}$ | 6500 | Paasivirta and Sinkkonen (2009) | V | |
| $C_{12}H_5Cl_5$ | $2.5\times10^{-2}$ | | Burkhard et al. (1985) | V | |
| (PCB-106) | $1.8\times10^{-1}$ | | Fang Lee (2007) | Q | 721 |
| [70424-69-0] | $1.4\times10^{-1}$ | | Fang Lee (2007) | Q | 722 |
| BQENMISTWGTJIJ-UHFFFAOYSA-N | $6.0\times10^{-2}$ | | Dunnivant et al. (1992) | Q | |
| | $5.1\times10^{-2}$ | | Sabljić and Güsten (1989) | Q | |



Table A6.3: Polychlorinated biphenyls (PCBs) (...continued)

| Substance / Formula / (Trivial Name) / [CAS Registry Number] / InChIKey | $H_s^{cp}$ (at $T^\ominus$) $\left[\dfrac{\text{mol}}{\text{m}^3\,\text{Pa}}\right]$ | $\dfrac{\text{d}\ln H_s^{cp}}{\text{d}(1/T)}$ [K] | Reference | Type | Note |
|---|---|---|---|---|---|
| 2,3,3',4,5-pentachlorobiphenyl | $4.3\times10^{-2}$ | 2200 | Bamford et al. (2002) | M | |
| $C_{12}H_5Cl_5$ | $1.7\times10^{-1}$ | | Murphy et al. (1987) | M | 12 |
| (PCB-107) | $9.1\times10^{-3}$ | 6200 | Paasivirta and Sinkkonen (2009) | V | |
| [70424-68-9] | $1.0\times10^{-1}$ | | Burkhard et al. (1985) | V | |
| PVYBHVJTMRRXLG-UHFFFAOYSA-N | $2.0\times10^{-1}$ | | Fang Lee (2007) | Q | 721 |
| | $1.2\times10^{-1}$ | | Fang Lee (2007) | Q | 722 |
| | $6.2\times10^{-2}$ | | Dunnivant et al. (1992) | Q | |
| | $4.9\times10^{-2}$ | | Sabljić and Güsten (1989) | Q | |
| 2,3,3',4,5'-pentachlorobiphenyl | $4.1\times10^{-3}$ | 5800 | Paasivirta and Sinkkonen (2009) | V | |
| $C_{12}H_5Cl_5$ | $1.0\times10^{-1}$ | | Burkhard et al. (1985) | V | |
| (PCB-108) | $1.1\times10^{-1}$ | | Fang Lee (2007) | Q | 721 |
| [70362-41-3] | $1.1\times10^{-1}$ | | Fang Lee (2007) | Q | 722 |
| MPCDNZSLJWJDNW-UHFFFAOYSA-N | $5.6\times10^{-2}$ | | Dunnivant et al. (1992) | Q | |
| | $2.8\times10^{-2}$ | | Sabljić and Güsten (1989) | Q | |
| 2,3,3',4,6-pentachlorobiphenyl | $1.5\times10^{-2}$ | 6600 | Paasivirta and Sinkkonen (2009) | V | |
| $C_{12}H_5Cl_5$ | $2.1\times10^{-2}$ | | Burkhard et al. (1985) | V | |
| (PCB-109) | $1.4\times10^{-1}$ | | Fang Lee (2007) | Q | 721 |
| [74472-35-8] | $1.2\times10^{-1}$ | | Fang Lee (2007) | Q | 722 |
| XGQBSVVYMVILEL-UHFFFAOYSA-N | $3.5\times10^{-2}$ | | Dunnivant et al. (1992) | Q | |
| | $2.5\times10^{-2}$ | | Sabljić and Güsten (1989) | Q | |
| 2,3,3',4',6-pentachlorobiphenyl | $2.3\times10^{-2}$ | 5200 | Bamford et al. (2002) | M | |
| $C_{12}H_5Cl_5$ | $9.3\times10^{-2}$ | | Murphy et al. (1987) | M | 12 |
| (PCB-110) | $2.7\times10^{-2}$ | | Murphy et al. (1983a) | M | 24 |
| [38380-03-9] | $1.8\times10^{-2}$ | 6400 | Paasivirta and Sinkkonen (2009) | V | |
| ARXHIJMGSIYYRZ-UHFFFAOYSA-N | $5.8\times10^{-2}$ | | Burkhard et al. (1985) | V | |
| | $8.3\times10^{-2}$ | | Fang Lee (2007) | Q | 721 |
| | $1.4\times10^{-1}$ | | Fang Lee (2007) | Q | 722 |
| | | 5000 | Kühne et al. (2005) | Q | |
| | $5.0\times10^{-2}$ | | Dunnivant et al. (1992) | Q | |
| | $5.2\times10^{-2}$ | | Sabljić and Güsten (1989) | Q | |
| | | 4300 | Kühne et al. (2005) | ? | |
| 2,3,3',5,5'-pentachlorobiphenyl | $6.5\times10^{-3}$ | 6200 | Paasivirta and Sinkkonen (2009) | V | |
| $C_{12}H_5Cl_5$ | $6.2\times10^{-2}$ | | Burkhard et al. (1985) | V | |
| (PCB-111) | $2.7\times10^{-1}$ | | Fang Lee (2007) | Q | 721 |
| [39635-32-0] | $7.8\times10^{-2}$ | | Fang Lee (2007) | Q | 722 |
| QMUDLTGWHILKHH-UHFFFAOYSA-N | $3.7\times10^{-2}$ | | Dunnivant et al. (1992) | Q | |
| | $2.0\times10^{-2}$ | | Sabljić and Güsten (1989) | Q | |
| 2,3,3',5,6-pentachlorobiphenyl | $7.8\times10^{-3}$ | 6600 | Paasivirta and Sinkkonen (2009) | V | |
| $C_{12}H_5Cl_5$ | $2.4\times10^{-2}$ | | Burkhard et al. (1985) | V | |
| (PCB-112) | $1.5\times10^{-1}$ | | Fang Lee (2007) | Q | 721 |
| [74472-36-9] | $2.0\times10^{-1}$ | | Fang Lee (2007) | Q | 722 |
| NTKSJAPQYKCFPP-UHFFFAOYSA-N | $3.7\times10^{-2}$ | | Dunnivant et al. (1992) | Q | |
| | $3.0\times10^{-2}$ | | Sabljić and Güsten (1989) | Q | |



Table A6.3: Polychlorinated biphenyls (PCBs) (...continued)

| Substance Formula (Trivial Name) [CAS Registry Number] InChIKey | $H_s^{cp}$ (at $T^{\ominus}$) $\left[\dfrac{\mathrm{mol}}{\mathrm{m}^3\,\mathrm{Pa}}\right]$ | $\dfrac{\mathrm{d}\ln H_s^{cp}}{\mathrm{d}(1/T)}$ [K] | Reference | Type | Note |
|---|---|---|---|---|---|
| 2,3,3',5',6-pentachlorobiphenyl | $1.5\times10^{-2}$ | 6500 | Paasivirta and Sinkkonen (2009) | V | |
| $C_{12}H_5Cl_5$ | $3.5\times10^{-2}$ | | Burkhard et al. (1985) | V | |
| (PCB-113) | $1.7\times10^{-1}$ | | Fang Lee (2007) | Q | 721 |
| [68194-10-5] | $1.2\times10^{-1}$ | | Fang Lee (2007) | Q | 722 |
| YDGFMDPEJCJZEV-UHFFFAOYSA-N | $3.0\times10^{-2}$ | | Dunnivant et al. (1992) | Q | |
| | $2.1\times10^{-2}$ | | Sabljić and Güsten (1989) | Q | |
| 2,3,4,4',5-pentachlorobiphenyl | $5.3\times10^{-2}$ | | Fang et al. (2006) | M | |
| $C_{12}H_5Cl_5$ | $1.4\times10^{-1}$ | | Murphy et al. (1987) | M | 12 |
| (PCB-114) | $1.2\times10^{-2}$ | 6400 | Paasivirta and Sinkkonen (2009) | V | |
| [74472-37-0] | $2.8\times10^{-2}$ | | Burkhard et al. (1985) | V | |
| SXZSFWHOSHAKMN-UHFFFAOYSA-N | $8.9\times10^{-2}$ | | Fang Lee (2007) | Q | 721 |
| | $1.3\times10^{-1}$ | | Fang Lee (2007) | Q | 722 |
| | $6.9\times10^{-2}$ | | Dunnivant et al. (1992) | Q | |
| | $8.7\times10^{-2}$ | | Sabljić and Güsten (1989) | Q | |
| 2,3,4,4',6-pentachlorobiphenyl | $2.6\times10^{-2}$ | 6900 | Paasivirta and Sinkkonen (2009) | V | |
| $C_{12}H_5Cl_5$ | $2.3\times10^{-2}$ | | Burkhard et al. (1985) | V | |
| (PCB-115) | $5.4\times10^{-2}$ | | Fang Lee (2007) | Q | 721 |
| [74472-38-1] | $1.1\times10^{-1}$ | | Fang Lee (2007) | Q | 722 |
| IOVARPVVZDOPGQ-UHFFFAOYSA-N | $4.0\times10^{-2}$ | | Dunnivant et al. (1992) | Q | |
| | $3.2\times10^{-2}$ | | Sabljić and Güsten (1989) | Q | |
| 2,3,4,5,6-pentachlorobiphenyl | $4.3\times10^{-3}$ | 6500 | Paasivirta and Sinkkonen (2009) | V | |
| $C_{12}H_5Cl_5$ | $5.5\times10^{-2}$ | | Burkhard et al. (1985) | V | |
| (PCB-116) | $9.9\times10^{-2}$ | | Fang Lee (2007) | Q | 721 |
| [18259-05-7] | $1.8\times10^{-1}$ | | Fang Lee (2007) | Q | 722 |
| GGMPTLAAIUQMIE-UHFFFAOYSA-N | $3.3\times10^{-2}$ | | Dunnivant et al. (1992) | Q | |
| | $4.3\times10^{-2}$ | | Sabljić and Güsten (1989) | Q | |
| 2,3,4',5,6-pentachlorobiphenyl | $1.5\times10^{-3}$ | 5900 | Paasivirta and Sinkkonen (2009) | V | |
| $C_{12}H_5Cl_5$ | $2.7\times10^{-2}$ | | Burkhard et al. (1985) | V | |
| (PCB-117) | $7.5\times10^{-2}$ | | Fang Lee (2007) | Q | 721 |
| [68194-11-6] | $1.7\times10^{-1}$ | | Fang Lee (2007) | Q | 722 |
| ZDDZPDTVCZLFFC-UHFFFAOYSA-N | $4.1\times10^{-2}$ | | Dunnivant et al. (1992) | Q | |
| | $4.0\times10^{-2}$ | | Sabljić and Güsten (1989) | Q | |
| 2,3',4,4',5-pentachlorobiphenyl | $3.1\times10^{-2}$ | 6800 | Li et al. (2003) | L | 366 |
| $C_{12}H_5Cl_5$ | $6.9\times10^{-2}$ | 7600 | Li et al. (2003) | L | 367 |
| (PCB-118) | $4.6\times10^{-3}$ | | Bhangare et al. (2019) | M | 725 |
| [31508-00-6] | $1.9\times10^{-2}$ | | Bhangare et al. (2019) | M | 726 |
| IUTPYMGCWINGEY-UHFFFAOYSA-N | $1.1\times10^{-2}$ | | Lau et al. (2006) | M | 719 |
| | $5.6\times10^{-3}$ | | Lau et al. (2006) | M | 720 |
| | $5.7\times10^{-2}$ | | Fang et al. (2006) | M | |
| | $1.8\times10^{-2}$ | 14000 | Charles and Destaillats (2005) | M | 33 |
| | $2.8\times10^{-2}$ | 6000 | Bamford et al. (2000) | M | |
| | $1.2\times10^{-1}$ | | Murphy et al. (1987) | M | 12 |
| | $2.5\times10^{-2}$ | | Murphy et al. (1983a) | M | 24 |
| | $3.4\times10^{-2}$ | | Duchowicz et al. (2020) | V | 186 |
| | $6.6\times10^{-3}$ | 6000 | Paasivirta and Sinkkonen (2009) | V | |



Table A6.3: Polychlorinated biphenyls (PCBs) (...continued)

| Substance<br>Formula<br>(Trivial Name)<br>[CAS Registry Number]<br>InChIKey | $H_s^{cp}$<br>(at $T^{\ominus}$)<br>$\left[\dfrac{\text{mol}}{\text{m}^3\,\text{Pa}}\right]$ | $\dfrac{\text{d}\ln H_s^{cp}}{\text{d}(1/T)}$<br><br>[K] | Reference | Type | Note |
|---|---|---|---|---|---|
|  | $1.1\times10^{-1}$ |  | Burkhard et al. (1985) | V |  |
|  | $2.6\times10^{-2}$ | 8100 | Paasivirta et al. (1999) | T |  |
|  | $1.5\times10^{-1}$ |  | Duchowicz et al. (2020) | Q |  |
|  | $7.8\times10^{-2}$ |  | Fang Lee (2007) | Q | 721 |
|  | $1.0\times10^{-1}$ |  | Fang Lee (2007) | Q | 722 |
|  |  | 5600 | Kühne et al. (2005) | Q |  |
|  | $7.9\times10^{-2}$ |  | Dunnivant et al. (1992) | Q |  |
|  | $8.5\times10^{-2}$ |  | Sabljić and Güsten (1989) | Q |  |
|  |  | 6300 | Kühne et al. (2005) | ? |  |
| 2,3',4,4',6-pentachlorobiphenyl<br>$C_{12}H_5Cl_5$<br>(PCB-119)<br>[56558-17-9]<br>OAEQTHQGPZKTQP-UHFFFAOYSA-N | $1.5\times10^{-2}$ | 4600 | Bamford et al. (2002) | M |  |
|  | $1.5\times10^{-2}$ | 6500 | Paasivirta and Sinkkonen (2009) | V |  |
|  | $4.4\times10^{-2}$ |  | Burkhard et al. (1985) | V |  |
|  | $2.4\times10^{-1}$ |  | Keshavarz et al. (2022) | Q |  |
|  | $9.4\times10^{-2}$ |  | Duchowicz et al. (2020) | Q | 184 |
|  | $1.6\times10^{-1}$ |  | Hilal et al. (2008) | Q |  |
|  | $1.4\times10^{-1}$ |  | Modarresi et al. (2007) | Q | 67 |
|  | $4.7\times10^{-2}$ |  | Fang Lee (2007) | Q | 721 |
|  | $7.4\times10^{-2}$ |  | Fang Lee (2007) | Q | 722 |
|  | $3.2\times10^{-2}$ |  | Dunnivant et al. (1992) | Q |  |
|  | $2.2\times10^{-2}$ |  | Sabljić and Güsten (1989) | Q |  |
|  | $1.3\times10^{-1}$ |  | Duchowicz et al. (2020) | ? | 185, 21 |
| 2,3',4,5,5'-pentachlorobiphenyl<br>$C_{12}H_5Cl_5$<br>(PCB-120)<br>[68194-12-7]<br>ZLGYJAIAVPVCNF-UHFFFAOYSA-N | $1.8\times10^{-1}$ |  | Brunner et al. (1990) | M |  |
|  | $3.9\times10^{-3}$ | 6000 | Paasivirta and Sinkkonen (2009) | V |  |
|  | $6.4\times10^{-2}$ |  | Burkhard et al. (1985) | V |  |
|  | $2.4\times10^{-1}$ |  | Keshavarz et al. (2022) | Q |  |
|  | $8.6\times10^{-2}$ |  | Duchowicz et al. (2020) | Q | 299 |
|  | $2.5\times10^{-1}$ |  | Hilal et al. (2008) | Q |  |
|  | $1.2\times10^{-1}$ |  | Modarresi et al. (2007) | Q | 67 |
|  | $1.6\times10^{-1}$ |  | Fang Lee (2007) | Q | 721 |
|  | $8.3\times10^{-2}$ |  | Fang Lee (2007) | Q | 722 |
|  | $4.0\times10^{-2}$ |  | Dunnivant et al. (1992) | Q |  |
|  | $2.4\times10^{-2}$ |  | Sabljić and Güsten (1989) | Q |  |
|  | $1.8\times10^{-1}$ |  | Duchowicz et al. (2020) | ? | 185, 21 |
| 2,3',4,5',6-pentachlorobiphenyl<br>$C_{12}H_5Cl_5$<br>(PCB-121)<br>[56558-18-0]<br>XBVSGJGMWSKAKL-UHFFFAOYSA-N | $8.5\times10^{-3}$ | 6500 | Paasivirta and Sinkkonen (2009) | V |  |
|  | $2.6\times10^{-2}$ |  | Burkhard et al. (1985) | V |  |
|  | $9.6\times10^{-2}$ |  | Fang Lee (2007) | Q | 721 |
|  | $6.2\times10^{-2}$ |  | Fang Lee (2007) | Q | 722 |
|  | $1.8\times10^{-2}$ |  | Dunnivant et al. (1992) | Q |  |
|  | $1.3\times10^{-2}$ |  | Sabljić and Güsten (1989) | Q |  |



Table A6.3: Polychlorinated biphenyls (PCBs) (...continued)

| Substance<br>Formula<br>(Trivial Name)<br>[CAS Registry Number]<br>InChIKey | $H_s^{cp}$<br>(at $T^{\ominus}$)<br>$\left[\dfrac{\mathrm{mol}}{\mathrm{m^3\,Pa}}\right]$ | $\dfrac{\mathrm{d}\ln H_s^{cp}}{\mathrm{d}(1/T)}$<br><br>[K] | Reference | Type | Note |
|---|---|---|---|---|---|
| 2,3,3',4',5'-pentachlorobiphenyl | $1.6\times10^{-1}$ | | Murphy et al. (1987) | M | 12 |
| $C_{12}H_5Cl_5$ | $4.3\times10^{-3}$ | 5800 | Paasivirta and Sinkkonen (2009) | V | |
| (PCB-122) | $1.6\times10^{-1}$ | | Burkhard et al. (1985) | V | |
| [76842-07-4] | $2.8\times10^{-1}$ | | Fang Lee (2007) | Q | 721 |
| GWOWBISZHLPYEK-UHFFFAOYSA-N | $1.4\times10^{-1}$ | | Fang Lee (2007) | Q | 722 |
| | $7.9\times10^{-2}$ | | Dunnivant et al. (1992) | Q | |
| | $7.2\times10^{-2}$ | | Sabljić and Güsten (1989) | Q | |
| 2,3',4,4',5'-pentachlorobiphenyl | $4.6\times10^{-2}$ | | Fang et al. (2006) | M | |
| $C_{12}H_5Cl_5$ | $3.7\times10^{-3}$ | 5800 | Paasivirta and Sinkkonen (2009) | V | |
| (PCB-123) | $1.1\times10^{-1}$ | | Burkhard et al. (1985) | V | |
| [65510-44-3] | $1.4\times10^{-1}$ | | Fang Lee (2007) | Q | 721 |
| YAHNWSSFXMVPOU-UHFFFAOYSA-N | $9.3\times10^{-2}$ | | Fang Lee (2007) | Q | 722 |
| | $5.7\times10^{-2}$ | | Dunnivant et al. (1992) | Q | |
| | $3.8\times10^{-2}$ | | Sabljić and Güsten (1989) | Q | |
| 2,3',4,5,5'-pentachlorobiphenyl | $1.9\times10^{-1}$ | | Murphy et al. (1987) | M | 12 |
| $C_{12}H_5Cl_5$ | $4.5\times10^{-3}$ | 5900 | Paasivirta and Sinkkonen (2009) | V | |
| (PCB-124) | $1.0\times10^{-1}$ | | Burkhard et al. (1985) | V | |
| [70424-70-3] | $2.8\times10^{-1}$ | | Fang Lee (2007) | Q | 721 |
| PIVBPZFQXKMHBD-UHFFFAOYSA-N | $9.4\times10^{-2}$ | | Fang Lee (2007) | Q | 722 |
| | $5.8\times10^{-2}$ | | Dunnivant et al. (1992) | Q | |
| | $5.1\times10^{-2}$ | | Sabljić and Güsten (1989) | Q | |
| 2,3',4,5',6-pentachlorobiphenyl | $2.3\times10^{-3}$ | 5800 | Paasivirta and Sinkkonen (2009) | V | |
| $C_{12}H_5Cl_5$ | $6.7\times10^{-2}$ | | Burkhard et al. (1985) | V | |
| (PCB-125) | $1.5\times10^{-1}$ | | Fang Lee (2007) | Q | 721 |
| [74472-39-2] | $1.1\times10^{-1}$ | | Fang Lee (2007) | Q | 722 |
| WAZUWHGJMMZVHH-UHFFFAOYSA-N | $3.4\times10^{-2}$ | | Dunnivant et al. (1992) | Q | |
| | $3.0\times10^{-2}$ | | Sabljić and Güsten (1989) | Q | |
| 3,3',4,4',5-pentachlorobiphenyl | $1.0\times10^{-1}$ | | Fang et al. (2006) | M | |
| $C_{12}H_5Cl_5$ | $4.8\times10^{-2}$ | 12000 | Bamford et al. (2000) | M | |
| (PCB-126) | $1.6\times10^{-3}$ | 5400 | Paasivirta and Sinkkonen (2009) | V | |
| [57465-28-8] | $3.6\times10^{-1}$ | | Burkhard et al. (1985) | V | |
| REHONNLQRWTIFF-UHFFFAOYSA-N | $6.5\times10^{-2}$ | 8800 | Paasivirta et al. (1999) | T | |
| | $1.8\times10^{-2}$ | | Bhangare et al. (2019) | Q | |
| | $1.7\times10^{-1}$ | | Fang Lee (2007) | Q | 721 |
| | $1.0\times10^{-1}$ | | Fang Lee (2007) | Q | 722 |
| | $1.2\times10^{-1}$ | | Dunnivant et al. (1992) | Q | |
| | $1.8\times10^{-1}$ | | Sabljić and Güsten (1989) | Q | |
| 3,3',4,5,5'-pentachlorobiphenyl | $2.2\times10^{-3}$ | 5600 | Paasivirta and Sinkkonen (2009) | V | |
| $C_{12}H_5Cl_5$ | $2.2\times10^{-1}$ | | Burkhard et al. (1985) | V | |
| (PCB-127) | $3.4\times10^{-1}$ | | Fang Lee (2007) | Q | 721 |
| [39635-33-1] | $8.4\times10^{-2}$ | | Fang Lee (2007) | Q | 722 |
| MXVAYAXIPRGORY-UHFFFAOYSA-N | $6.3\times10^{-2}$ | | Dunnivant et al. (1992) | Q | |
| | $2.9\times10^{-2}$ | | Sabljić and Güsten (1989) | Q | |





Table A6.3: Polychlorinated biphenyls (PCBs) (...continued)

| Substance<br>Formula<br>(Trivial Name)<br>[CAS Registry Number]<br>InChIKey | $H_s^{cp}$<br>(at $T^{\ominus}$)<br>$\left[\dfrac{\mathrm{mol}}{\mathrm{m}^3\,\mathrm{Pa}}\right]$ | $\dfrac{\mathrm{d}\ln H_s^{cp}}{\mathrm{d}(1/T)}$<br><br>[K] | Reference | Type | Note |
|---|---|---|---|---|---|
| 2,2',3,3',4,4'-hexachlorobiphenyl | $2.8\times10^{-2}$ | 14000 | Bamford et al. (2000) | M | |
| $C_{12}H_4Cl_6$ | $7.6\times10^{-1}$ | | Brunner et al. (1990) | M | |
| (PCB-128) | $3.3\times10^{-1}$ | | Dunnivant et al. (1988) | M | |
| [38380-07-3] | $3.3\times10^{-1}$ | | Dunnivant and Elzerman (1988) | M | 723 |
| BTAGRXWGMYTPBY-UHFFFAOYSA-N | $1.7\times10^{-1}$ | | Murphy et al. (1987) | M | 12 |
| | $2.0\times10^{-2}$ | | Murphy et al. (1983a) | M | 24 |
| | $6.9\times10^{-3}$ | 6100 | Paasivirta and Sinkkonen (2009) | V | |
| | $8.4\times10^{-2}$ | | Mackay et al. (2006b) | V | |
| | $8.4\times10^{-2}$ | | Mackay et al. (1992a) | V | |
| | $8.3\times10^{-2}$ | | Shiu and Mackay (1986) | V | |
| | $1.5\times10^{-1}$ | | Burkhard et al. (1985) | V | |
| | $2.0\times10^{-2}$ | | Murphy et al. (1983b) | X | 724, 24 |
| | $2.4\times10^{-1}$ | | Keshavarz et al. (2022) | Q | |
| | $3.8\times10^{-1}$ | | Duchowicz et al. (2020) | Q | 299 |
| | $1.5\times10^{-2}$ | | Bhangare et al. (2019) | Q | |
| | $1.8\times10^{-1}$ | | Hilal et al. (2008) | Q | |
| | $2.1\times10^{-1}$ | | Modarresi et al. (2007) | Q | 67 |
| | $1.5\times10^{-1}$ | | Fang Lee (2007) | Q | 721 |
| | $2.4\times10^{-1}$ | | Fang Lee (2007) | Q | 722 |
| | $1.2\times10^{-1}$ | | English and Carroll (2001) | Q | 230, 231 |
| | $9.5\times10^{-2}$ | | Dunnivant et al. (1992) | Q | |
| | $7.6\times10^{-1}$ | | Duchowicz et al. (2020) | ? | 185, 21 |
| 2,2',3,3',4,5-hexachlorobiphenyl | $3.4\times10^{-1}$ | | Brunner et al. (1990) | M | |
| $C_{12}H_4Cl_6$ | $6.4\times10^{-3}$ | 6400 | Paasivirta and Sinkkonen (2009) | V | |
| (PCB-129) | $2.5\times10^{-2}$ | | Burkhard et al. (1985) | V | |
| [55215-18-4] | $2.4\times10^{-1}$ | | Keshavarz et al. (2022) | Q | |
| VQQKIXKPMJTUMP-UHFFFAOYSA-N | $2.7\times10^{-1}$ | | Duchowicz et al. (2020) | Q | |
| | $1.6\times10^{-1}$ | | Hilal et al. (2008) | Q | |
| | $2.2\times10^{-1}$ | | Modarresi et al. (2007) | Q | 67 |
| | $2.9\times10^{-1}$ | | Fang Lee (2007) | Q | 721 |
| | $2.6\times10^{-1}$ | | Fang Lee (2007) | Q | 722 |
| | $7.1\times10^{-2}$ | | Dunnivant et al. (1992) | Q | |
| | $1.2\times10^{-1}$ | | Sabljić and Güsten (1989) | Q | |
| | $3.4\times10^{-1}$ | | Duchowicz et al. (2020) | ? | 185, 21 |
| 2,2',3,3',4,5'-hexachlorobiphenyl | $2.7\times10^{-1}$ | | Brunner et al. (1990) | M | |
| $C_{12}H_4Cl_6$ | $9.2\times10^{-2}$ | | Murphy et al. (1987) | M | 12 |
| (PCB-130) | $5.9\times10^{-3}$ | 6500 | Paasivirta and Sinkkonen (2009) | V | |
| [52663-66-8] | $8.7\times10^{-2}$ | | Burkhard et al. (1985) | V | |
| YFSLABAYQDPWPF-UHFFFAOYSA-N | $2.4\times10^{-1}$ | | Keshavarz et al. (2022) | Q | |
| | $2.1\times10^{-1}$ | | Duchowicz et al. (2020) | Q | |
| | $1.9\times10^{-1}$ | | Hilal et al. (2008) | Q | |
| | $2.0\times10^{-1}$ | | Modarresi et al. (2007) | Q | 67 |
| | $3.1\times10^{-1}$ | | Fang Lee (2007) | Q | 721 |
| | $2.1\times10^{-1}$ | | Fang Lee (2007) | Q | 722 |
| | $6.5\times10^{-2}$ | | Dunnivant et al. (1992) | Q | |
| | $5.1\times10^{-2}$ | | Sabljić and Güsten (1989) | Q | |



Table A6.3: Polychlorinated biphenyls (PCBs) (... continued)

| Substance Formula (Trivial Name) [CAS Registry Number] InChIKey | $H_s^{cp}$ (at $T^{\ominus}$) $\left[\dfrac{\text{mol}}{\text{m}^3\,\text{Pa}}\right]$ | $\dfrac{\text{d}\ln H_s^{cp}}{\text{d}(1/T)}$ [K] | Reference | Type | Note |
|---|---|---|---|---|---|
| | $2.7\times10^{-1}$ | | Duchowicz et al. (2020) | ? | 185, 21 |
| 2,2',3,3',4,6-hexachlorobiphenyl | $1.5\times10^{-1}$ | | Murphy et al. (1987) | M | 12 |
| $C_{12}H_4Cl_6$ | $3.0\times10^{-3}$ | 6500 | Paasivirta and Sinkkonen (2009) | V | |
| (PCB-131) | $1.6\times10^{-2}$ | | Burkhard et al. (1985) | V | |
| [61798-70-7] | $2.4\times10^{-1}$ | | Keshavarz et al. (2022) | Q | |
| WDLTVNWWEZJMPF-UHFFFAOYSA-N | $1.6\times10^{-1}$ | | Duchowicz et al. (2020) | Q | 299 |
| | $1.6\times10^{-1}$ | | Hilal et al. (2008) | Q | |
| | $2.1\times10^{-1}$ | | Modarresi et al. (2007) | Q | 67 |
| | $1.7\times10^{-1}$ | | Fang Lee (2007) | Q | 721 |
| | $1.9\times10^{-1}$ | | Fang Lee (2007) | Q | 722 |
| | $4.1\times10^{-2}$ | | Dunnivant et al. (1992) | Q | |
| | $3.8\times10^{-2}$ | | Sabljić and Güsten (1989) | Q | |
| | $2.5\times10^{-1}$ | | Duchowicz et al. (2020) | ? | 185, 21 |
| 2,2',3,3',4,6'-hexachlorobiphenyl | $4.0\times10^{-2}$ | 2400 | Bamford et al. (2002) | M | |
| $C_{12}H_4Cl_6$ | $2.2\times10^{-1}$ | | Brunner et al. (1990) | M | |
| (PCB-132) | $4.1\times10^{-3}$ | 6400 | Paasivirta and Sinkkonen (2009) | V | |
| [38380-05-1] | $3.6\times10^{-2}$ | | Burkhard et al. (1985) | V | |
| OKBJVIVEFXPEOU-UHFFFAOYSA-N | $2.4\times10^{-1}$ | | Keshavarz et al. (2022) | Q | |
| | $2.1\times10^{-1}$ | | Duchowicz et al. (2020) | Q | 299 |
| | $2.2\times10^{-1}$ | | Hilal et al. (2008) | Q | |
| | $1.6\times10^{-1}$ | | Modarresi et al. (2007) | Q | 67 |
| | $1.7\times10^{-1}$ | | Fang Lee (2007) | Q | 721 |
| | $2.1\times10^{-1}$ | | Fang Lee (2007) | Q | 722 |
| | $2.3\times10^{-1}$ | | Yaffe et al. (2003) | Q | 248, 249 |
| | $4.9\times10^{-2}$ | | Dunnivant et al. (1992) | Q | |
| | $6.1\times10^{-2}$ | | Sabljić and Güsten (1989) | Q | |
| | $2.2\times10^{-1}$ | | Duchowicz et al. (2020) | ? | 185, 21 |
| 2,2',3,3',5,5'-hexachlorobiphenyl | $4.1\times10^{-3}$ | 6500 | Paasivirta and Sinkkonen (2009) | V | |
| $C_{12}H_4Cl_6$ | $5.2\times10^{-2}$ | | Burkhard et al. (1985) | V | |
| (PCB-133) | $1.8\times10^{-1}$ | | Hilal et al. (2008) | Q | |
| [35694-04-3] | $4.3\times10^{-1}$ | | Fang Lee (2007) | Q | 721 |
| AJKLKINFZLWHQE-UHFFFAOYSA-N | $2.0\times10^{-1}$ | | Fang Lee (2007) | Q | 722 |
| | $4.8\times10^{-2}$ | | Dunnivant et al. (1992) | Q | |
| | $3.0\times10^{-2}$ | | Sabljić and Güsten (1989) | Q | |
| 2,2',3,3',5,6-hexachlorobiphenyl | $1.2\times10^{-2}$ | 7300 | Bamford et al. (2002) | M | |
| $C_{12}H_4Cl_6$ | $2.0\times10^{-1}$ | | Brunner et al. (1990) | M | |
| (PCB-134) | $1.0\times10^{-1}$ | | Murphy et al. (1987) | M | 12 |
| [52704-70-8] | $1.7\times10^{-2}$ | | Murphy et al. (1983a) | M | 24 |
| RVWLHPJFOKUPNM-UHFFFAOYSA-N | $2.8\times10^{-3}$ | 6400 | Paasivirta and Sinkkonen (2009) | V | |
| | $1.8\times10^{-2}$ | | Burkhard et al. (1985) | V | |
| | $2.4\times10^{-1}$ | | Keshavarz et al. (2022) | Q | |
| | $1.5\times10^{-1}$ | | Duchowicz et al. (2020) | Q | 299 |
| | $2.0\times10^{-1}$ | | Hilal et al. (2008) | Q | |
| | $2.1\times10^{-1}$ | | Modarresi et al. (2007) | Q | 67 |
| | $2.4\times10^{-1}$ | | Fang Lee (2007) | Q | 721 |





Table A6.3: Polychlorinated biphenyls (PCBs) (. . . continued)

| Substance Formula (Trivial Name) [CAS Registry Number] InChIKey | $H_s^{cp}$ (at $T^{\ominus}$) $\left[\dfrac{\mathrm{mol}}{\mathrm{m^3\,Pa}}\right]$ | $\dfrac{\mathrm{d}\ln H_s^{cp}}{\mathrm{d}(1/T)}$ [K] | Reference | Type | Note |
|---|---|---|---|---|---|
| | $3.2\times10^{-1}$ | | Fang Lee (2007) | Q | 722 |
| | $4.3\times10^{-2}$ | | Dunnivant et al. (1992) | Q | |
| | $4.9\times10^{-2}$ | | Sabljić and Güsten (1989) | Q | |
| | $2.0\times10^{-1}$ | | Duchowicz et al. (2020) | ? | 185, 21 |
| 2,2',3,3',5,6'-hexachlorobiphenyl | $1.5\times10^{-2}$ | 5500 | Bamford et al. (2002) | M | |
| $C_{12}H_4Cl_6$ | $1.8\times10^{-1}$ | | Brunner et al. (1990) | M | |
| (PCB-135) | $7.0\times10^{-2}$ | | Murphy et al. (1987) | M | 12 |
| [52744-13-5] | $4.8\times10^{-3}$ | 6600 | Paasivirta and Sinkkonen (2009) | V | |
| UUTNFLRSJBQQJM-UHFFFAOYSA-N | $2.1\times10^{-2}$ | | Burkhard et al. (1985) | V | |
| | $2.4\times10^{-1}$ | | Keshavarz et al. (2022) | Q | |
| | $1.1\times10^{-1}$ | | Duchowicz et al. (2020) | Q | 184 |
| | $2.3\times10^{-1}$ | | Hilal et al. (2008) | Q | |
| | $1.9\times10^{-1}$ | | Modarresi et al. (2007) | Q | 67 |
| | $2.4\times10^{-1}$ | | Fang Lee (2007) | Q | 721 |
| | $2.1\times10^{-1}$ | | Fang Lee (2007) | Q | 722 |
| | $3.7\times10^{-2}$ | | Dunnivant et al. (1992) | Q | |
| | $3.2\times10^{-2}$ | | Sabljić and Güsten (1989) | Q | |
| | $1.8\times10^{-1}$ | | Duchowicz et al. (2020) | ? | 185, 21 |
| 2,2',3,3',6,6'-hexachlorobiphenyl | $9.0\times10^{-3}$ | 5400 | Bamford et al. (2002) | M | |
| $C_{12}H_4Cl_6$ | $1.1\times10^{-1}$ | | Brunner et al. (1990) | M | |
| (PCB-136) | $4.4\times10^{-2}$ | | Murphy et al. (1987) | M | 12 |
| [38411-22-2] | $1.6\times10^{-3}$ | 6300 | Paasivirta and Sinkkonen (2009) | V | |
| FZFUUSROAHKTTF-UHFFFAOYSA-N | $1.1\times10^{-2}$ | | Burkhard et al. (1985) | V | |
| | $2.4\times10^{-1}$ | | Keshavarz et al. (2022) | Q | |
| | $1.1\times10^{-1}$ | | Duchowicz et al. (2020) | Q | 299 |
| | $2.7\times10^{-1}$ | | Hilal et al. (2008) | Q | |
| | $1.5\times10^{-1}$ | | Modarresi et al. (2007) | Q | 67 |
| | $1.4\times10^{-1}$ | | Fang Lee (2007) | Q | 721 |
| | $1.6\times10^{-1}$ | | Fang Lee (2007) | Q | 722 |
| | $3.1\times10^{-2}$ | | Dunnivant et al. (1992) | Q | |
| | $3.9\times10^{-2}$ | | Sabljić and Güsten (1989) | Q | |
| | $1.1\times10^{-1}$ | | Duchowicz et al. (2020) | ? | 185, 21 |
| 2,2',3,4,4',5-hexachlorobiphenyl | $4.5\times10^{-2}$ | 3200 | Bamford et al. (2002) | M | |
| $C_{12}H_4Cl_6$ | $1.5\times10^{-1}$ | | Murphy et al. (1987) | M | 12 |
| (PCB-137) | $2.1\times10^{-2}$ | | Murphy et al. (1983a) | M | 24 |
| [35694-06-5] | $1.8\times10^{-2}$ | 6800 | Paasivirta and Sinkkonen (2009) | V | |
| CKLLRBPBZLTGDJ-UHFFFAOYSA-N | $1.8\times10^{-2}$ | | Burkhard et al. (1985) | V | |
| | $1.4\times10^{-1}$ | | Fang Lee (2007) | Q | 721 |
| | $1.7\times10^{-1}$ | | Fang Lee (2007) | Q | 722 |
| | $5.3\times10^{-2}$ | | Dunnivant et al. (1992) | Q | |
| | $4.7\times10^{-2}$ | | Sabljić and Güsten (1989) | Q | |





Table A6.3: Polychlorinated biphenyls (PCBs) (...continued)

| Substance Formula (Trivial Name) [CAS Registry Number] InChIKey | $H_s^{cp}$ (at $T^{\ominus}$) $\left[\dfrac{\mathrm{mol}}{\mathrm{m^3\,Pa}}\right]$ | $\dfrac{\mathrm{d\ln} H_s^{cp}}{\mathrm{d}(1/T)}$ [K] | Reference | Type | Note |
|---|---|---|---|---|---|
| 2,2',3,4,4',5'-hexachlorobiphenyl | $2.5\times10^{-2}$ | 7100 | Li et al. (2003) | L | 366 |
| $C_{12}H_4Cl_6$ | $3.3\times10^{-2}$ | 7700 | Li et al. (2003) | L | 367 |
| (PCB-138) | $2.2\times10^{-2}$ | 10000 | Bamford et al. (2000) | M | |
| [35065-28-2] | $4.7\times10^{-1}$ | | Brunner et al. (1990) | M | |
| RPUMZMSNLZHIGZ-UHFFFAOYSA-N | $1.3\times10^{-1}$ | | Murphy et al. (1987) | M | 12 |
| | $1.4\times10^{-1}$ | | Brownawell (1986) | M | 294 |
| | $2.1\times10^{-2}$ | | Murphy et al. (1983a) | M | 24 |
| | $1.8\times10^{-2}$ | 6800 | Paasivirta and Sinkkonen (2009) | V | |
| | $1.2\times10^{-2}$ | | Shiu and Mackay (1986) | V | |
| | $9.1\times10^{-2}$ | | Burkhard et al. (1985) | V | |
| | $4.7\times10^{-2}$ | 8700 | Paasivirta et al. (1999) | T | |
| | $2.4\times10^{-1}$ | | Keshavarz et al. (2022) | Q | |
| | $2.1\times10^{-1}$ | | Duchowicz et al. (2020) | Q | 184 |
| | $1.5\times10^{-2}$ | | Bhangare et al. (2019) | Q | |
| | $1.8\times10^{-1}$ | | Hilal et al. (2008) | Q | |
| | $2.2\times10^{-1}$ | | Modarresi et al. (2007) | Q | 67 |
| | $1.6\times10^{-1}$ | | Fang Lee (2007) | Q | 721 |
| | $1.8\times10^{-1}$ | | Fang Lee (2007) | Q | 722 |
| | $5.2\times10^{-1}$ | | Yaffe et al. (2003) | Q | 248, 249 |
| | $7.6\times10^{-2}$ | | Dunnivant et al. (1992) | Q | |
| | $9.2\times10^{-2}$ | | Sabljić and Güsten (1989) | Q | |
| | $4.7\times10^{-1}$ | | Duchowicz et al. (2020) | ? | 185, 21 |
| 2,2',3,4,4',6-hexachlorobiphenyl | $1.4\times10^{-2}$ | 6900 | Paasivirta and Sinkkonen (2009) | V | |
| $C_{12}H_4Cl_6$ | $1.1\times10^{-2}$ | | Burkhard et al. (1985) | V | |
| (PCB-139) | $8.6\times10^{-2}$ | | Fang Lee (2007) | Q | 721 |
| [56030-56-9] | $1.3\times10^{-1}$ | | Fang Lee (2007) | Q | 722 |
| SPOPSCCFZQFGDL-UHFFFAOYSA-N | $3.0\times10^{-2}$ | | Dunnivant et al. (1992) | Q | |
| | $2.6\times10^{-2}$ | | Sabljić and Güsten (1989) | Q | |
| 2,2',3,4,4',6'-hexachlorobiphenyl | $1.7\times10^{-2}$ | 7000 | Paasivirta and Sinkkonen (2009) | V | |
| $C_{12}H_4Cl_6$ | $2.7\times10^{-2}$ | | Burkhard et al. (1985) | V | |
| (PCB-140) | $8.5\times10^{-2}$ | | Fang Lee (2007) | Q | 721 |
| [59291-64-4] | $1.1\times10^{-1}$ | | Fang Lee (2007) | Q | 722 |
| XBBRGUHRZBZMPP-UHFFFAOYSA-N | $3.2\times10^{-2}$ | | Dunnivant et al. (1992) | Q | |
| | $2.3\times10^{-2}$ | | Sabljić and Güsten (1989) | Q | |
| 2,2',3,4,5,5'-hexachlorobiphenyl | $2.0\times10^{-2}$ | 8400 | Bamford et al. (2002) | M | |
| $C_{12}H_4Cl_6$ | $4.3\times10^{-1}$ | | Brunner et al. (1990) | M | |
| (PCB-141) | $1.0\times10^{-1}$ | | Murphy et al. (1987) | M | 12 |
| [52712-04-6] | $2.5\times10^{-2}$ | | Murphy et al. (1983a) | M | 24 |
| UCLKLGIYGBLTSM-UHFFFAOYSA-N | $1.0\times10^{-2}$ | 6700 | Paasivirta and Sinkkonen (2009) | V | |
| | $2.5\times10^{-2}$ | | Shiu and Mackay (1986) | V | |
| | $1.6\times10^{-2}$ | | Burkhard et al. (1985) | V | |
| | $2.4\times10^{-1}$ | | Keshavarz et al. (2022) | Q | |
| | $1.5\times10^{-1}$ | | Duchowicz et al. (2020) | Q | 184 |
| | $1.3\times10^{-1}$ | | Hilal et al. (2008) | Q | |
| | $2.0\times10^{-1}$ | | Modarresi et al. (2007) | Q | 67 |
| | $2.9\times10^{-1}$ | | Fang Lee (2007) | Q | 721 |





Table A6.3: Polychlorinated biphenyls (PCBs) (...continued)

| Substance<br>Formula<br>(Trivial Name)<br>[CAS Registry Number]<br>InChIKey | $H_s^{cp}$<br>(at $T^\ominus$)<br>$\left[\dfrac{\mathrm{mol}}{\mathrm{m^3\,Pa}}\right]$ | $\dfrac{\mathrm{d}\ln H_s^{cp}}{\mathrm{d}(1/T)}$<br><br>[K] | Reference | Type | Note |
|---|---|---|---|---|---|
| | $1.8\times10^{-1}$ | | Fang Lee (2007) | Q | 722 |
| | $5.7\times10^{-2}$ | | Dunnivant et al. (1992) | Q | |
| | $6.9\times10^{-2}$ | | Sabljić and Güsten (1989) | Q | |
| | $4.3\times10^{-1}$ | | Duchowicz et al. (2020) | ? | 185, 21 |
| 2,2',3,4,5,6-hexachlorobiphenyl | $4.0\times10^{-3}$ | 6900 | Paasivirta and Sinkkonen (2009) | V | |
| $C_{12}H_4Cl_6$ | $1.4\times10^{-2}$ | | Burkhard et al. (1985) | V | |
| (PCB-142) | $1.6\times10^{-1}$ | | Fang Lee (2007) | Q | 721 |
| [41411-61-4] | $2.4\times10^{-1}$ | | Fang Lee (2007) | Q | 722 |
| RUEIBQJFGMERJD-UHFFFAOYSA-N | $3.1\times10^{-2}$ | | Dunnivant et al. (1992) | Q | |
| | $4.7\times10^{-2}$ | | Sabljić and Güsten (1989) | Q | |
| 2,2',3,4,5,6'-hexachlorobiphenyl | $6.5\times10^{-3}$ | 6600 | Paasivirta and Sinkkonen (2009) | V | |
| $C_{12}H_4Cl_6$ | $7.8\times10^{-3}$ | | Burkhard et al. (1985) | V | |
| (PCB-143) | $2.4\times10^{-1}$ | | Keshavarz et al. (2022) | Q | |
| [68194-15-0] | $1.6\times10^{-1}$ | | Duchowicz et al. (2020) | Q | 184 |
| UQPQKLGBEKEZBV-UHFFFAOYSA-N | $1.6\times10^{-1}$ | | Hilal et al. (2008) | Q | |
| | $1.7\times10^{-1}$ | | Modarresi et al. (2007) | Q | 67 |
| | $1.6\times10^{-1}$ | | Fang Lee (2007) | Q | 721 |
| | $1.9\times10^{-1}$ | | Fang Lee (2007) | Q | 722 |
| | $3.4\times10^{-2}$ | | Dunnivant et al. (1992) | Q | |
| | $3.9\times10^{-2}$ | | Sabljić and Güsten (1989) | Q | |
| | $2.5\times10^{-1}$ | | Duchowicz et al. (2020) | ? | 185, 21 |
| 2,2',3,4,5',6-hexachlorobiphenyl | $7.0\times10^{-2}$ | | Murphy et al. (1987) | M | 12 |
| $C_{12}H_4Cl_6$ | $1.6\times10^{-2}$ | | Murphy et al. (1983a) | M | 24 |
| (PCB-144) | $1.2\times10^{-2}$ | 7000 | Paasivirta and Sinkkonen (2009) | V | |
| [68194-14-9] | $1.7\times10^{-2}$ | | Shiu and Mackay (1986) | V | |
| CXXRQFOKRZJAJA-UHFFFAOYSA-N | $1.0\times10^{-2}$ | | Burkhard et al. (1985) | V | |
| | $1.8\times10^{-1}$ | | Fang Lee (2007) | Q | 721 |
| | $1.4\times10^{-1}$ | | Fang Lee (2007) | Q | 722 |
| | $3.3\times10^{-2}$ | | Dunnivant et al. (1992) | Q | |
| | $3.1\times10^{-2}$ | | Sabljić and Güsten (1989) | Q | |
| 2,2',3,4,6,6'-hexachlorobiphenyl | $1.5\times10^{-3}$ | 6400 | Paasivirta and Sinkkonen (2009) | V | |
| $C_{12}H_4Cl_6$ | $5.9\times10^{-3}$ | | Burkhard et al. (1985) | V | |
| (PCB-145) | $9.6\times10^{-2}$ | | Fang Lee (2007) | Q | 721 |
| [74472-40-5] | $1.1\times10^{-1}$ | | Fang Lee (2007) | Q | 722 |
| JZFZCLFEPXCRCA-UHFFFAOYSA-N | $2.1\times10^{-2}$ | | Dunnivant et al. (1992) | Q | |
| | $2.4\times10^{-2}$ | | Sabljić and Güsten (1989) | Q | |
| 2,2',3,4',5,5'-hexachlorobiphenyl | $1.7\times10^{-2}$ | 7100 | Bamford et al. (2002) | M | |
| $C_{12}H_4Cl_6$ | $3.9\times10^{-1}$ | | Brunner et al. (1990) | M | |
| (PCB-146) | $1.1\times10^{-1}$ | | Murphy et al. (1987) | M | 12 |
| [51908-16-8] | $1.2\times10^{-2}$ | 6800 | Paasivirta and Sinkkonen (2009) | V | |
| BQHCQAQLTCQFJZ-UHFFFAOYSA-N | $5.4\times10^{-2}$ | | Burkhard et al. (1985) | V | |
| | $2.4\times10^{-1}$ | | Keshavarz et al. (2022) | Q | |
| | $1.1\times10^{-1}$ | | Duchowicz et al. (2020) | Q | 184 |
| | $2.0\times10^{-1}$ | | Hilal et al. (2008) | Q | |
| | $2.2\times10^{-1}$ | | Modarresi et al. (2007) | Q | 67 |





Table A6.3: Polychlorinated biphenyls (PCBs) (...continued)

| Substance / Formula / (Trivial Name) / [CAS Registry Number] / InChIKey | $H_s^{cp}$ (at $T^{\ominus}$) $\left[\dfrac{\mathrm{mol}}{\mathrm{m^3\,Pa}}\right]$ | $\dfrac{\mathrm{d}\ln H_s^{cp}}{\mathrm{d}(1/T)}$ [K] | Reference | Type | Note |
|---|---|---|---|---|---|
| | $2.2\times10^{-1}$ | | Fang Lee (2007) | Q | 721 |
| | $1.6\times10^{-1}$ | | Fang Lee (2007) | Q | 722 |
| | $5.3\times10^{-2}$ | | Dunnivant et al. (1992) | Q | |
| | $4.0\times10^{-2}$ | | Sabljić and Güsten (1989) | Q | |
| | $3.9\times10^{-1}$ | | Duchowicz et al. (2020) | ? | 185, 21 |
| 2,2',3,4',5,6-hexachlorobiphenyl | $1.9\times10^{-1}$ | | Brunner et al. (1990) | M | |
| $C_{12}H_4Cl_6$ | $3.1\times10^{-3}$ | 6500 | Paasivirta and Sinkkonen (2009) | V | |
| (PCB-147) | $1.3\times10^{-2}$ | | Burkhard et al. (1985) | V | |
| [68194-13-8] | $2.4\times10^{-1}$ | | Keshavarz et al. (2022) | Q | |
| AQONCPKMJSBHQT-UHFFFAOYSA-N | $8.7\times10^{-2}$ | | Duchowicz et al. (2020) | Q | 299 |
| | $1.8\times10^{-1}$ | | Hilal et al. (2008) | Q | |
| | $2.1\times10^{-1}$ | | Modarresi et al. (2007) | Q | 67 |
| | $1.2\times10^{-1}$ | | Fang Lee (2007) | Q | 721 |
| | $2.1\times10^{-1}$ | | Fang Lee (2007) | Q | 722 |
| | $3.1\times10^{-2}$ | | Dunnivant et al. (1992) | Q | |
| | $3.1\times10^{-2}$ | | Sabljić and Güsten (1989) | Q | |
| | $1.9\times10^{-1}$ | | Duchowicz et al. (2020) | ? | 185, 21 |
| 2,2',3,4',5,6'-hexachlorobiphenyl | $1.2\times10^{-2}$ | 7000 | Paasivirta and Sinkkonen (2009) | V | |
| $C_{12}H_4Cl_6$ | $1.6\times10^{-2}$ | | Burkhard et al. (1985) | V | |
| (PCB-148) | $1.2\times10^{-1}$ | | Fang Lee (2007) | Q | 721 |
| [74472-41-6] | $1.0\times10^{-1}$ | | Fang Lee (2007) | Q | 722 |
| CTVRBEKNQHJPLX-UHFFFAOYSA-N | $2.3\times10^{-2}$ | | Dunnivant et al. (1992) | Q | |
| | $1.7\times10^{-2}$ | | Sabljić and Güsten (1989) | Q | |
| 2,2',3,4',5',6-hexachlorobiphenyl | $1.5\times10^{-2}$ | 5500 | Bamford et al. (2002) | M | |
| $C_{12}H_4Cl_6$ | $6.7\times10^{-2}$ | | Murphy et al. (1987) | M | 12 |
| (PCB-149) | $3.3\times10^{-2}$ | | Murphy et al. (1983a) | M | 24 |
| [38380-04-0] | $7.2\times10^{-3}$ | | Duchowicz et al. (2020) | V | 186 |
| LKHLFUVHHXCNJH-UHFFFAOYSA-N | $1.0\times10^{-2}$ | 6800 | Paasivirta and Sinkkonen (2009) | V | |
| | $3.3\times10^{-2}$ | | Shiu and Mackay (1986) | V | |
| | $2.2\times10^{-2}$ | | Burkhard et al. (1985) | V | |
| | $1.1\times10^{-1}$ | | Duchowicz et al. (2020) | Q | |
| | $1.3\times10^{-1}$ | | Fang Lee (2007) | Q | 721 |
| | $1.7\times10^{-1}$ | | Fang Lee (2007) | Q | 722 |
| | $2.1\times10^{-1}$ | | Yaffe et al. (2003) | Q | 248, 249 |
| | $4.2\times10^{-2}$ | | Dunnivant et al. (1992) | Q | |
| | $4.5\times10^{-2}$ | | Sabljić and Güsten (1989) | Q | |
| 2,2',3,4',6,6'-hexachlorobiphenyl | $6.5\times10^{-3}$ | 6900 | Paasivirta and Sinkkonen (2009) | V | |
| $C_{12}H_4Cl_6$ | $8.2\times10^{-3}$ | | Burkhard et al. (1985) | V | |
| (PCB-150) | $7.2\times10^{-2}$ | | Fang Lee (2007) | Q | 721 |
| [68194-08-1] | $8.0\times10^{-2}$ | | Fang Lee (2007) | Q | 722 |
| RPPNJBZNXQNKNM-UHFFFAOYSA-N | $2.0\times10^{-2}$ | | Dunnivant et al. (1992) | Q | |
| | $1.9\times10^{-2}$ | | Sabljić and Güsten (1989) | Q | |



Table A6.3: Polychlorinated biphenyls (PCBs) (. . . continued)

| Substance<br>Formula<br>(Trivial Name)<br>[CAS Registry Number]<br>InChIKey | $H_s^{cp}$<br>(at $T^{\ominus}$)<br>$\left[\dfrac{\text{mol}}{\text{m}^3\,\text{Pa}}\right]$ | $\dfrac{\text{d}\ln H_s^{cp}}{\text{d}(1/T)}$<br>[K] | Reference | Type | Note |
|---|---|---|---|---|---|
| 2,2',3,5,5',6-hexachlorobiphenyl | $1.4\times10^{-2}$ | 4500 | Bamford et al. (2002) | M | |
| $C_{12}H_4Cl_6$ | $1.7\times10^{-1}$ | | Brunner et al. (1990) | M | |
| (PCB-151) | $6.3\times10^{-2}$ | | Murphy et al. (1987) | M | 12 |
| [52663-63-5] | $3.3\times10^{-2}$ | | Murphy et al. (1983a) | M | 24 |
| UHCLFIWDCYOTOL-UHFFFAOYSA-N | $5.2\times10^{-3}$ | 6700 | Paasivirta and Sinkkonen (2009) | V | |
| | $3.3\times10^{-2}$ | | Shiu and Mackay (1986) | V | |
| | $1.2\times10^{-2}$ | | Burkhard et al. (1985) | V | |
| | $2.4\times10^{-1}$ | | Keshavarz et al. (2022) | Q | |
| | $7.9\times10^{-2}$ | | Duchowicz et al. (2020) | Q | |
| | $1.6\times10^{-1}$ | | Hilal et al. (2008) | Q | |
| | $2.1\times10^{-1}$ | | Modarresi et al. (2007) | Q | 67 |
| | $2.4\times10^{-1}$ | | Fang Lee (2007) | Q | 721 |
| | $2.4\times10^{-1}$ | | Fang Lee (2007) | Q | 722 |
| | $3.5\times10^{-2}$ | | Dunnivant et al. (1992) | Q | |
| | $3.8\times10^{-2}$ | | Sabljić and Güsten (1989) | Q | |
| | $1.7\times10^{-1}$ | | Duchowicz et al. (2020) | ? | 185, 21 |
| 2,2',3,5,6,6'-hexachlorobiphenyl | $1.3\times10^{-3}$ | 6300 | Paasivirta and Sinkkonen (2009) | V | |
| $C_{12}H_4Cl_6$ | $6.8\times10^{-3}$ | | Burkhard et al. (1985) | V | |
| (PCB-152) | $1.3\times10^{-1}$ | | Fang Lee (2007) | Q | 721 |
| [68194-09-2] | $1.9\times10^{-1}$ | | Fang Lee (2007) | Q | 722 |
| CLODVDBWNVQLGO-UHFFFAOYSA-N | $2.3\times10^{-2}$ | | Dunnivant et al. (1992) | Q | |
| | $2.8\times10^{-2}$ | | Sabljić and Güsten (1989) | Q | |
| 2,2',4,4',5,5'-hexachlorobiphenyl | $4.0\times10^{-2}$ | 7100 | Li et al. (2003) | L | 366 |
| $C_{12}H_4Cl_6$ | $5.1\times10^{-2}$ | 7900 | Li et al. (2003) | L | 367 |
| (PCB-153) | $1.9\times10^{-2}$ | 8000 | Bamford et al. (2000) | M | |
| [35065-27-1] | $4.3\times10^{-1}$ | | Brunner et al. (1990) | M | |
| MVWHGTYKUMDIHL-UHFFFAOYSA-N | $7.5\times10^{-2}$ | | Dunnivant et al. (1988) | M | |
| | $7.5\times10^{-2}$ | | Dunnivant and Elzerman (1988) | M | 723 |
| | $1.0\times10^{-1}$ | | Murphy et al. (1987) | M | 12 |
| | $1.6\times10^{-1}$ | | Oliver (1985) | M | |
| | $2.8\times10^{-2}$ | | Murphy et al. (1983a) | M | 24 |
| | $1.1\times10^{-2}$ | 6700 | Paasivirta and Sinkkonen (2009) | V | |
| | $2.3\times10^{-2}$ | | Mackay et al. (2006b) | V | |
| | $2.3\times10^{-2}$ | | Mackay et al. (1992a) | V | |
| | $2.3\times10^{-2}$ | | Shiu and Mackay (1986) | V | |
| | $5.6\times10^{-2}$ | | Burkhard et al. (1985) | V | |
| | $1.7\times10^{-2}$ | 8400 | Paasivirta et al. (1999) | T | |
| | $2.8\times10^{-2}$ | | Murphy et al. (1983b) | X | 724, 24 |
| | $8.0\times10^{-2}$ | | Dunnivant et al. (1988) | C | |
| | $2.4\times10^{-1}$ | | Keshavarz et al. (2022) | Q | |
| | $1.1\times10^{-1}$ | | Duchowicz et al. (2020) | Q | 299 |
| | $1.7\times10^{-2}$ | | Bhangare et al. (2019) | Q | |
| | $1.8\times10^{-1}$ | | Hilal et al. (2008) | Q | |
| | $2.4\times10^{-1}$ | | Modarresi et al. (2007) | Q | 67 |
| | $1.2\times10^{-1}$ | | Fang Lee (2007) | Q | 721 |
| | $1.4\times10^{-1}$ | | Fang Lee (2007) | Q | 722 |



Table A6.3: Polychlorinated biphenyls (PCBs) (...continued)

| Substance Formula (Trivial Name) [CAS Registry Number] InChIKey | $H_s^{cp}$ (at $T^{\ominus}$) $\left[\dfrac{\mathrm{mol}}{\mathrm{m^3\,Pa}}\right]$ | $\dfrac{\mathrm{d}\ln H_s^{cp}}{\mathrm{d}(1/T)}$ [K] | Reference | Type | Note |
|---|---|---|---|---|---|
| | $2.9\times10^{-1}$ | | Yaffe et al. (2003) | Q | 248, 249 |
| | $6.0\times10^{-2}$ | | Dunnivant et al. (1992) | Q | |
| | $4.3\times10^{-1}$ | | Duchowicz et al. (2020) | ? | 185, 21 |
| 2,2',4,4',5,6'-hexachlorobiphenyl | $1.3\times10^{-2}$ | 5600 | Bamford et al. (2000) | M | |
| $C_{12}H_4Cl_6$ | $1.7\times10^{-2}$ | 7100 | Paasivirta and Sinkkonen (2009) | V | |
| (PCB-154) | $1.7\times10^{-2}$ | | Burkhard et al. (1985) | V | |
| [60145-22-4] | $1.8\times10^{-2}$ | | Bhangare et al. (2019) | Q | |
| QXZHEJWDLVUFFB-UHFFFAOYSA-N | $6.8\times10^{-2}$ | | Fang Lee (2007) | Q | 721 |
| | $8.7\times10^{-2}$ | | Fang Lee (2007) | Q | 722 |
| | $2.6\times10^{-2}$ | | Dunnivant et al. (1992) | Q | |
| | $2.0\times10^{-2}$ | | Sabljić and Güsten (1989) | Q | |
| 2,2',4,4',6,6'-hexachlorobiphenyl | $1.3\times10^{-2}$ | 7100 | Li et al. (2003) | L | 366 |
| $C_{12}H_4Cl_6$ | $1.1\times10^{-2}$ | 7600 | Li et al. (2003) | L | 367 |
| (PCB-155) | $1.3\times10^{-2}$ | | Dunnivant et al. (1988) | M | |
| [33979-03-2] | $1.3\times10^{-2}$ | | Dunnivant and Elzerman (1988) | M | 723 |
| ICOAEPDGFWLUTI-UHFFFAOYSA-N | $3.9\times10^{-3}$ | | Duchowicz et al. (2020) | V | 186 |
| | $4.6\times10^{-3}$ | 6900 | Paasivirta and Sinkkonen (2009) | V | |
| | $1.2\times10^{-2}$ | | Mackay et al. (2006b) | V | |
| | $1.2\times10^{-2}$ | | Mackay et al. (1992a) | V | |
| | $1.2\times10^{-3}$ | | Shiu and Mackay (1986) | V | |
| | $6.4\times10^{-3}$ | | Burkhard et al. (1985) | V | |
| | $8.6\times10^{-2}$ | | Dunnivant et al. (1988) | C | |
| | $4.8\times10^{-2}$ | | Duchowicz et al. (2020) | Q | |
| | $4.2\times10^{-2}$ | | Fang Lee (2007) | Q | 721 |
| | $4.2\times10^{-2}$ | | Fang Lee (2007) | Q | 722 |
| | $1.2\times10^{-2}$ | | Dunnivant et al. (1992) | Q | |
| 2,3,3',4,4',5-hexachlorobiphenyl | $6.8\times10^{-2}$ | | Fang et al. (2006) | M | |
| $C_{12}H_4Cl_6$ | $2.9\times10^{-2}$ | 13000 | Bamford et al. (2002) | M | |
| (PCB-156) | $1.1\times10^{-2}$ | | Murphy et al. (1983a) | M | 24 |
| [38380-08-4] | $6.9\times10^{-2}$ | | Duchowicz et al. (2020) | V | 186 |
| LCXMEXLGMKFLQO-UHFFFAOYSA-N | $5.9\times10^{-3}$ | 6200 | Paasivirta and Sinkkonen (2009) | V | |
| | $1.1\times10^{-2}$ | | Shiu and Mackay (1986) | V | |
| | $5.7\times10^{-2}$ | | Burkhard et al. (1985) | V | |
| | $2.7\times10^{-1}$ | | Duchowicz et al. (2020) | Q | |
| | $1.8\times10^{-1}$ | | Fang Lee (2007) | Q | 721 |
| | $2.0\times10^{-1}$ | | Fang Lee (2007) | Q | 722 |
| | $1.1\times10^{-1}$ | | Dunnivant et al. (1992) | Q | |
| | $4.5\times10^{-1}$ | | Sabljić and Güsten (1989) | Q | |
| 2,3,3',4,4',5'-hexachlorobiphenyl | $6.0\times10^{-2}$ | | Fang et al. (2006) | M | |
| $C_{12}H_4Cl_6$ | $3.4\times10^{-2}$ | 16000 | Bamford et al. (2002) | M | |
| (PCB-157) | $1.7\times10^{-2}$ | | Murphy et al. (1983a) | M | 24 |
| [69782-90-7] | $2.3\times10^{-3}$ | 5900 | Paasivirta and Sinkkonen (2009) | V | |
| YTWXDQVNPCIEOX-UHFFFAOYSA-N | $1.7\times10^{-2}$ | | Shiu and Mackay (1986) | V | |
| | $3.0\times10^{-1}$ | | Burkhard et al. (1985) | V | |
| | $1.9\times10^{-1}$ | | Fang Lee (2007) | Q | 721 |





Table A6.3: Polychlorinated biphenyls (PCBs) (. . . continued)

| Substance Formula (Trivial Name) [CAS Registry Number] InChIKey | $H_s^{cp}$ (at $T^\ominus$) $\left[\dfrac{\text{mol}}{\text{m}^3\,\text{Pa}}\right]$ | $\dfrac{\text{d}\ln H_s^{cp}}{\text{d}(1/T)}$ [K] | Reference | Type | Note |
|---|---|---|---|---|---|
| | $2.0\times10^{-1}$ | | Fang Lee (2007) | Q | 722 |
| | | 6300 | Kühne et al. (2005) | Q | |
| | $1.2\times10^{-1}$ | | Dunnivant et al. (1992) | Q | |
| | $1.5\times10^{-1}$ | | Sabljić and Güsten (1989) | Q | |
| | | 5100 | Kühne et al. (2005) | ? | |
| 2,3,3',4,4',6-hexachlorobiphenyl | $2.1\times10^{-2}$ | 9600 | Bamford et al. (2002) | M | |
| $C_{12}H_4Cl_6$ | $2.3\times10^{-1}$ | | Murphy et al. (1987) | M | 12 |
| (PCB-158) | $1.5\times10^{-2}$ | | Murphy et al. (1983a) | M | 24 |
| [74472-42-7] | $9.2\times10^{-3}$ | 6600 | Paasivirta and Sinkkonen (2009) | V | |
| ZQUPQXINXTWCQR-UHFFFAOYSA-N | $1.5\times10^{-2}$ | | Shiu and Mackay (1986) | V | |
| | $4.8\times10^{-2}$ | | Burkhard et al. (1985) | V | |
| | $1.1\times10^{-1}$ | | Fang Lee (2007) | Q | 721 |
| | $1.9\times10^{-1}$ | | Fang Lee (2007) | Q | 722 |
| | $6.0\times10^{-2}$ | | Dunnivant et al. (1992) | Q | |
| | $4.6\times10^{-2}$ | | Sabljić and Güsten (1989) | Q | |
| 2,3,3',4,5,5'-hexachlorobiphenyl | $4.9\times10^{-1}$ | | Brunner et al. (1990) | M | |
| $C_{12}H_4Cl_6$ | $2.9\times10^{-3}$ | 6100 | Paasivirta and Sinkkonen (2009) | V | |
| (PCB-159) | $3.4\times10^{-2}$ | | Burkhard et al. (1985) | V | |
| [39635-35-3] | $2.4\times10^{-1}$ | | Keshavarz et al. (2022) | Q | |
| YZKLGRAIIIGOHB-UHFFFAOYSA-N | $1.6\times10^{-1}$ | | Duchowicz et al. (2020) | Q | 184 |
| | $2.6\times10^{-1}$ | | Hilal et al. (2008) | Q | |
| | $1.9\times10^{-1}$ | | Modarresi et al. (2007) | Q | 67 |
| | $3.6\times10^{-1}$ | | Fang Lee (2007) | Q | 721 |
| | $1.8\times10^{-1}$ | | Fang Lee (2007) | Q | 722 |
| | $6.3\times10^{-2}$ | | Dunnivant et al. (1992) | Q | |
| | $3.2\times10^{-2}$ | | Sabljić and Güsten (1989) | Q | |
| | $4.9\times10^{-1}$ | | Duchowicz et al. (2020) | ? | 185, 21 |
| 2,3,3',4,5,6-hexachlorobiphenyl | $4.9\times10^{-1}$ | | Brunner et al. (1990) | M | |
| $C_{12}H_4Cl_6$ | $7.9\times10^{-3}$ | 7100 | Paasivirta and Sinkkonen (2009) | V | |
| (PCB-160) | $4.0\times10^{-2}$ | | Burkhard et al. (1985) | V | |
| [41411-62-5] | $2.4\times10^{-1}$ | | Keshavarz et al. (2022) | Q | |
| JHJMZCXLJXRCHK-UHFFFAOYSA-N | $1.1\times10^{-1}$ | | Duchowicz et al. (2020) | Q | 184 |
| | $1.2\times10^{-1}$ | | Hilal et al. (2008) | Q | |
| | $3.8\times10^{-1}$ | | Modarresi et al. (2007) | Q | 67 |
| | $2.0\times10^{-1}$ | | Fang Lee (2007) | Q | 721 |
| | $3.3\times10^{-1}$ | | Fang Lee (2007) | Q | 722 |
| | $4.6\times10^{-2}$ | | Dunnivant et al. (1992) | Q | |
| | $3.9\times10^{-2}$ | | Sabljić and Güsten (1989) | Q | |
| | $4.9\times10^{-1}$ | | Duchowicz et al. (2020) | ? | 185, 21 |
| 2,3,3',4,5',6-hexachlorobiphenyl | $9.2\times10^{-3}$ | 6800 | Paasivirta and Sinkkonen (2009) | V | |
| $C_{12}H_4Cl_6$ | $2.9\times10^{-2}$ | | Burkhard et al. (1985) | V | |
| (PCB-161) | $2.2\times10^{-1}$ | | Fang Lee (2007) | Q | 721 |
| [74472-43-8] | $1.7\times10^{-1}$ | | Fang Lee (2007) | Q | 722 |
| UNPTZXSJGZTGJJ-UHFFFAOYSA-N | $3.5\times10^{-2}$ | | Dunnivant et al. (1992) | Q | |
| | $2.0\times10^{-2}$ | | Sabljić and Güsten (1989) | Q | |



Table A6.3: Polychlorinated biphenyls (PCBs) (. . . continued)

| Substance Formula (Trivial Name) [CAS Registry Number] InChIKey | $H_s^{cp}$ (at $T^{\ominus}$) $\left[\dfrac{\mathrm{mol}}{\mathrm{m^3\,Pa}}\right]$ | $\dfrac{\mathrm{d}\ln H_s^{cp}}{\mathrm{d}(1/T)}$ [K] | Reference | Type | Note |
|---|---|---|---|---|---|
| 2,3,3',4',5,5'-hexachlorobiphenyl | $3.7\times10^{-3}$ | 6200 | Paasivirta and Sinkkonen (2009) | V | |
| $C_{12}H_4Cl_6$ | $1.8\times10^{-1}$ | | Burkhard et al. (1985) | V | |
| (PCB-162) | $2.7\times10^{-1}$ | | Fang Lee (2007) | Q | 721 |
| [39635-34-2] | $1.8\times10^{-1}$ | | Fang Lee (2007) | Q | 722 |
| AHZUOPSGLWYCNF-UHFFFAOYSA-N | $7.5\times10^{-2}$ | | Dunnivant et al. (1992) | Q | |
| | $4.8\times10^{-2}$ | | Sabljić and Güsten (1989) | Q | |
| 2,3,3',4',5,6-hexachlorobiphenyl | $2.1\times10^{-2}$ | 9700 | Bamford et al. (2002) | M | |
| $C_{12}H_4Cl_6$ | $6.6\times10^{-1}$ | | Brunner et al. (1990) | M | |
| (PCB-163) | $4.9\times10^{-3}$ | 6400 | Paasivirta and Sinkkonen (2009) | V | |
| [74472-44-9] | $5.4\times10^{-2}$ | | Burkhard et al. (1985) | V | |
| ZAGRQXMWMRUYRB-UHFFFAOYSA-N | $2.4\times10^{-1}$ | | Keshavarz et al. (2022) | Q | |
| | $1.5\times10^{-1}$ | | Duchowicz et al. (2020) | Q | 299 |
| | $2.5\times10^{-1}$ | | Hilal et al. (2008) | Q | |
| | $2.6\times10^{-1}$ | | Modarresi et al. (2007) | Q | 67 |
| | $1.5\times10^{-1}$ | | Fang Lee (2007) | Q | 721 |
| | $3.1\times10^{-1}$ | | Fang Lee (2007) | Q | 722 |
| | $6.0\times10^{-2}$ | | Dunnivant et al. (1992) | Q | |
| | $6.3\times10^{-2}$ | | Sabljić and Güsten (1989) | Q | |
| | $6.6\times10^{-1}$ | | Duchowicz et al. (2020) | ? | 185, 21 |
| 2,3,3',4',5',6-hexachlorobiphenyl | $9.5\times10^{-3}$ | 6500 | Paasivirta and Sinkkonen (2009) | V | |
| $C_{12}H_4Cl_6$ | $1.0\times10^{-1}$ | | Burkhard et al. (1985) | V | |
| (PCB-164) | $1.6\times10^{-1}$ | | Fang Lee (2007) | Q | 721 |
| [74472-45-0] | $2.3\times10^{-1}$ | | Fang Lee (2007) | Q | 722 |
| HAZQOLYHFUUJJN-UHFFFAOYSA-N | $5.6\times10^{-2}$ | | Dunnivant et al. (1992) | Q | |
| | $5.0\times10^{-2}$ | | Sabljić and Güsten (1989) | Q | |
| 2,3,3',5,5',6-hexachlorobiphenyl | $3.4\times10^{-1}$ | | Brunner et al. (1990) | M | |
| $C_{12}H_4Cl_6$ | $2.5\times10^{-3}$ | 6400 | Paasivirta and Sinkkonen (2009) | V | |
| (PCB-165) | $3.2\times10^{-2}$ | | Burkhard et al. (1985) | V | |
| [74472-46-1] | $2.4\times10^{-1}$ | | Keshavarz et al. (2022) | Q | |
| ZEATXTCWXKQPHO-UHFFFAOYSA-N | $8.7\times10^{-2}$ | | Duchowicz et al. (2020) | Q | |
| | $1.6\times10^{-1}$ | | Hilal et al. (2008) | Q | |
| | $2.1\times10^{-1}$ | | Modarresi et al. (2007) | Q | 67 |
| | $3.0\times10^{-1}$ | | Fang Lee (2007) | Q | 721 |
| | $2.9\times10^{-1}$ | | Fang Lee (2007) | Q | 722 |
| | $3.6\times10^{-2}$ | | Dunnivant et al. (1992) | Q | |
| | $2.2\times10^{-2}$ | | Sabljić and Güsten (1989) | Q | |
| | $3.4\times10^{-1}$ | | Duchowicz et al. (2020) | ? | 185, 21 |
| 2,3,4,4',5,6-hexachlorobiphenyl | $3.5\times10^{-3}$ | 6500 | Paasivirta and Sinkkonen (2009) | V | |
| $C_{12}H_4Cl_6$ | $4.4\times10^{-2}$ | | Burkhard et al. (1985) | V | |
| (PCB-166) | $1.2\times10^{-1}$ | | Hilal et al. (2008) | Q | |
| [41411-63-6] | $3.1\times10^{-1}$ | | Modarresi et al. (2007) | Q | 67 |
| BTOCFTAWZMMTNB-UHFFFAOYSA-N | $9.8\times10^{-2}$ | | Fang Lee (2007) | Q | 721 |
| | $2.9\times10^{-1}$ | | Fang Lee (2007) | Q | 722 |
| | | 4100 | Kühne et al. (2005) | Q | |
| | $5.4\times10^{-2}$ | | Dunnivant et al. (1992) | Q | |





Table A6.3: Polychlorinated biphenyls (PCBs) (. . . continued)

| Substance Formula (Trivial Name) [CAS Registry Number] InChIKey | $H_s^{cp}$ (at $T^\ominus$) $\left[\dfrac{\text{mol}}{\text{m}^3\,\text{Pa}}\right]$ | $\dfrac{\text{d}\ln H_s^{cp}}{\text{d}(1/T)}$ [K] | Reference | Type | Note |
|---|---|---|---|---|---|
| | $5.7\times10^{-2}$ | | Sabljić and Güsten (1989) | Q | |
| | | 5800 | Kühne et al. (2005) | ? | |
| 2,3',4,4',5,5'-hexachlorobiphenyl | $7.8\times10^{-2}$ | | Fang et al. (2006) | M | |
| $C_{12}H_4Cl_6$ | $2.7\times10^{-2}$ | 13000 | Bamford et al. (2002) | M | |
| (PCB-167) | $7.3\times10^{-3}$ | 6400 | Paasivirta and Sinkkonen (2009) | V | |
| [52663-72-6] | $1.9\times10^{-1}$ | | Burkhard et al. (1985) | V | |
| AZXHAWRMEPZSSV-UHFFFAOYSA-N | $1.6\times10^{-1}$ | | Fang Lee (2007) | Q | 721 |
| | $1.4\times10^{-1}$ | | Fang Lee (2007) | Q | 722 |
| | $9.0\times10^{-2}$ | | Dunnivant et al. (1992) | Q | |
| | $8.0\times10^{-2}$ | | Sabljić and Güsten (1989) | Q | |
| 2,3',4,4',5',6-hexachlorobiphenyl | $1.1\times10^{-2}$ | 6700 | Paasivirta and Sinkkonen (2009) | V | |
| $C_{12}H_4Cl_6$ | $7.7\times10^{-2}$ | | Burkhard et al. (1985) | V | |
| (PCB-168) | $9.4\times10^{-2}$ | | Fang Lee (2007) | Q | 721 |
| [59291-65-5] | $1.2\times10^{-1}$ | | Fang Lee (2007) | Q | 722 |
| PITHIPNORFGJPI-UHFFFAOYSA-N | $3.6\times10^{-2}$ | | Dunnivant et al. (1992) | Q | |
| | $2.1\times10^{-2}$ | | Sabljić and Güsten (1989) | Q | |
| 3,3',4,4',5,5'-hexachlorobiphenyl | $8.1\times10^{-2}$ | | Fang et al. (2006) | M | |
| $C_{12}H_4Cl_6$ | $4.7\times10^{-2}$ | 19000 | Bamford et al. (2002) | M | |
| (PCB-169) | $4.0\times10^{-4}$ | 5100 | Paasivirta and Sinkkonen (2009) | V | |
| [32774-16-6] | $6.4\times10^{-1}$ | | Burkhard et al. (1985) | V | |
| ZHLICBPIXDOFFG-UHFFFAOYSA-N | $2.3\times10^{-2}$ | 9000 | Paasivirta et al. (1999) | T | |
| | $3.4\times10^{-1}$ | | Fang Lee (2007) | Q | 721 |
| | $1.3\times10^{-1}$ | | Fang Lee (2007) | Q | 722 |
| | $1.5\times10^{-1}$ | | Dunnivant et al. (1992) | Q | |
| | $1.7\times10^{-1}$ | | Sabljić and Güsten (1989) | Q | |
| 2,2',3,3',4,4',5-heptachlorobiphenyl | $4.8\times10^{-2}$ | 20000 | Bamford et al. (2000) | M | |
| $C_{12}H_3Cl_7$ | 1.1 | | Brunner et al. (1990) | M | |
| (PCB-170) | $6.6\times10^{-1}$ | | Murphy et al. (1987) | M | 12 |
| [35065-30-6] | $7.8\times10^{-3}$ | 6600 | Paasivirta and Sinkkonen (2009) | V | |
| RMPWIIKNWPVWNG-UHFFFAOYSA-N | $5.2\times10^{-2}$ | | Burkhard et al. (1985) | V | |
| | $2.4\times10^{-1}$ | | Keshavarz et al. (2022) | Q | |
| | $3.9\times10^{-1}$ | | Duchowicz et al. (2020) | Q | 184 |
| | $1.5\times10^{-2}$ | | Bhangare et al. (2019) | Q | |
| | $2.1\times10^{-1}$ | | Hilal et al. (2008) | Q | |
| | $3.2\times10^{-1}$ | | Modarresi et al. (2007) | Q | 67 |
| | $2.8\times10^{-1}$ | | Fang Lee (2007) | Q | 721 |
| | $4.0\times10^{-1}$ | | Fang Lee (2007) | Q | 722 |
| | $1.1\times10^{-1}$ | | Dunnivant et al. (1992) | Q | |
| | 1.1 | | Duchowicz et al. (2020) | ? | 185, 21 |



Table A6.3: Polychlorinated biphenyls (PCBs) (...continued)

| Substance Formula (Trivial Name) [CAS Registry Number] InChIKey | $H_s^{cp}$ (at $T^{\ominus}$) $\left[\dfrac{\mathrm{mol}}{\mathrm{m}^3\,\mathrm{Pa}}\right]$ | $\dfrac{\mathrm{d}\ln H_s^{cp}}{\mathrm{d}(1/T)}$ [K] | Reference | Type | Note |
|---|---|---|---|---|---|
| 2,2',3,3',4,4',6-heptachlorobiphenyl | $2.9\times10^{-2}$ | | Duchowicz et al. (2020) | V | 186 |
| $C_{12}H_3Cl_7$ | $1.3\times10^{-2}$ | 7100 | Paasivirta and Sinkkonen (2009) | V | |
| (PCB-171) | $1.9\times10^{-1}$ | | Mackay et al. (2006b) | V | |
| [52663-71-5] | $1.9\times10^{-1}$ | | Mackay et al. (1992a) | V | |
| TZMHVHLTPWKZCI-UHFFFAOYSA-N | $1.9\times10^{-1}$ | | Shiu and Mackay (1986) | V | |
| | $3.4\times10^{-2}$ | | Burkhard et al. (1985) | V | |
| | $2.3\times10^{-1}$ | | Duchowicz et al. (2020) | Q | |
| | $2.1\times10^{-1}$ | | Hilal et al. (2008) | Q | |
| | $2.8\times10^{-1}$ | | Modarresi et al. (2007) | Q | 67 |
| | $1.7\times10^{-1}$ | | Fang Lee (2007) | Q | 721 |
| | $3.1\times10^{-1}$ | | Fang Lee (2007) | Q | 722 |
| | $5.7\times10^{-2}$ | | Dunnivant et al. (1992) | Q | |
| 2,2',3,3',4,5,5'-heptachlorobiphenyl | $7.6\times10^{-1}$ | | Brunner et al. (1990) | M | |
| $C_{12}H_3Cl_7$ | $8.3\times10^{-3}$ | 6800 | Paasivirta and Sinkkonen (2009) | V | |
| (PCB-172) | $3.1\times10^{-2}$ | | Burkhard et al. (1985) | V | |
| [52663-74-8] | $2.4\times10^{-1}$ | | Keshavarz et al. (2022) | Q | |
| HOPMUCXYRNOABF-UHFFFAOYSA-N | $2.1\times10^{-1}$ | | Duchowicz et al. (2020) | Q | 184 |
| | $3.2\times10^{-1}$ | | Modarresi et al. (2007) | Q | 67 |
| | $5.6\times10^{-1}$ | | Fang Lee (2007) | Q | 721 |
| | $3.8\times10^{-1}$ | | Fang Lee (2007) | Q | 722 |
| | $8.3\times10^{-2}$ | | Dunnivant et al. (1992) | Q | |
| | 7.6 | | Duchowicz et al. (2020) | ? | 185, 21 |
| 2,2',3,3',4,5,6-heptachlorobiphenyl | $7.0\times10^{-1}$ | | Brunner et al. (1990) | M | |
| $C_{12}H_3Cl_7$ | $1.3\times10^{-3}$ | 6500 | Paasivirta and Sinkkonen (2009) | V | |
| (PCB-173) | $3.3\times10^{-2}$ | | Burkhard et al. (1985) | V | |
| [68194-16-1] | $2.4\times10^{-1}$ | | Keshavarz et al. (2022) | Q | |
| PAYFWJAKKLILIT-UHFFFAOYSA-N | $2.1\times10^{-1}$ | | Duchowicz et al. (2020) | Q | |
| | $1.8\times10^{-1}$ | | Hilal et al. (2008) | Q | |
| | $3.5\times10^{-1}$ | | Modarresi et al. (2007) | Q | 67 |
| | $3.1\times10^{-1}$ | | Fang Lee (2007) | Q | 721 |
| | $5.9\times10^{-1}$ | | Fang Lee (2007) | Q | 722 |
| | $5.4\times10^{-2}$ | | Dunnivant et al. (1992) | Q | |
| | $7.0\times10^{-1}$ | | Duchowicz et al. (2020) | ? | 185, 21 |
| 2,2',3,3',4,5,6'-heptachlorobiphenyl | $2.2\times10^{-2}$ | 14000 | Bamford et al. (2002) | M | |
| $C_{12}H_3Cl_7$ | $7.0\times10^{-1}$ | | Brunner et al. (1990) | M | |
| (PCB-174) | $2.0\times10^{-1}$ | | Murphy et al. (1987) | M | 12 |
| [38411-25-5] | $5.4\times10^{-3}$ | 6700 | Paasivirta and Sinkkonen (2009) | V | |
| ZDLMBNHYTPHDLF-UHFFFAOYSA-N | $1.3\times10^{-2}$ | | Burkhard et al. (1985) | V | |
| | $2.4\times10^{-1}$ | | Keshavarz et al. (2022) | Q | |
| | $2.1\times10^{-1}$ | | Duchowicz et al. (2020) | Q | 299 |
| | $2.6\times10^{-1}$ | | Hilal et al. (2008) | Q | |
| | $3.3\times10^{-1}$ | | Modarresi et al. (2007) | Q | 67 |
| | $3.1\times10^{-1}$ | | Fang Lee (2007) | Q | 721 |





Table A6.3: Polychlorinated biphenyls (PCBs) (...continued)

| Substance Formula (Trivial Name) [CAS Registry Number] InChIKey | $H_s^{cp}$ (at $T^\ominus$) $\left[\dfrac{\mathrm{mol}}{\mathrm{m^3\,Pa}}\right]$ | $\dfrac{\mathrm{d}\ln H_s^{cp}}{\mathrm{d}(1/T)}$ [K] | Reference | Type | Note |
|---|---|---|---|---|---|
| | $4.3\times10^{-1}$ | | Fang Lee (2007) | Q | 722 |
| | $5.8\times10^{-2}$ | | Dunnivant et al. (1992) | Q | |
| | $7.0\times10^{-1}$ | | Duchowicz et al. (2020) | ? | 185, 21 |
| 2,2',3,3',4,5',6-heptachlorobiphenyl | $1.0\times10^{-2}$ | 7200 | Paasivirta and Sinkkonen (2009) | V | |
| $C_{12}H_3Cl_7$ | $2.0\times10^{-2}$ | | Burkhard et al. (1985) | V | |
| (PCB-175) | $3.4\times10^{-1}$ | | Fang Lee (2007) | Q | 721 |
| [40186-70-7] | $3.0\times10^{-1}$ | | Fang Lee (2007) | Q | 722 |
| KJBDZJFSYQUNJT-UHFFFAOYSA-N | $4.4\times10^{-2}$ | | Dunnivant et al. (1992) | Q | |
| 2,2',3,3',4,6,6'-heptachlorobiphenyl | $1.1\times10^{-1}$ | | Murphy et al. (1987) | M | 12 |
| $C_{12}H_3Cl_7$ | $8.5\times10^{-3}$ | 7200 | Paasivirta and Sinkkonen (2009) | V | |
| (PCB-176) | $1.0\times10^{-2}$ | | Burkhard et al. (1985) | V | |
| [52663-65-7] | $1.9\times10^{-1}$ | | Fang Lee (2007) | Q | 721 |
| HGMYRFJAJNYBRX-UHFFFAOYSA-N | $2.6\times10^{-1}$ | | Fang Lee (2007) | Q | 722 |
| | $3.3\times10^{-2}$ | | Dunnivant et al. (1992) | Q | |
| 2,2',3,3',4,5',6'-heptachlorobiphenyl | $2.1\times10^{-2}$ | 13000 | Bamford et al. (2002) | M | |
| $C_{12}H_3Cl_7$ | $3.0\times10^{-1}$ | | Murphy et al. (1987) | M | 12 |
| (PCB-177) | $4.5\times10^{-3}$ | | Murphy et al. (1983a) | M | 24 |
| [52663-70-4] | $3.4\times10^{-3}$ | 6600 | Paasivirta and Sinkkonen (2009) | V | |
| CXOYNJAHPUASHN-UHFFFAOYSA-N | $3.8\times10^{-2}$ | | Burkhard et al. (1985) | V | |
| | $2.4\times10^{-1}$ | | Fang Lee (2007) | Q | 721 |
| | $5.3\times10^{-1}$ | | Fang Lee (2007) | Q | 722 |
| | $6.0\times10^{-2}$ | | Dunnivant et al. (1992) | Q | |
| 2,2',3,3',5,5',6-heptachlorobiphenyl | $1.5\times10^{-2}$ | 11000 | Bamford et al. (2002) | M | |
| $C_{12}H_3Cl_7$ | $4.3\times10^{-1}$ | | Brunner et al. (1990) | M | |
| (PCB-178) | $1.5\times10^{-1}$ | | Murphy et al. (1987) | M | 12 |
| [52663-67-9] | $1.0\times10^{-2}$ | 7200 | Paasivirta and Sinkkonen (2009) | V | |
| WCIBKXHMIXUQHK-UHFFFAOYSA-N | $2.3\times10^{-2}$ | | Burkhard et al. (1985) | V | |
| | $2.4\times10^{-1}$ | | Keshavarz et al. (2022) | Q | |
| | $1.2\times10^{-1}$ | | Duchowicz et al. (2020) | Q | 184 |
| | $2.7\times10^{-1}$ | | Hilal et al. (2008) | Q | |
| | $3.0\times10^{-1}$ | | Modarresi et al. (2007) | Q | 67 |
| | $4.8\times10^{-1}$ | | Fang Lee (2007) | Q | 721 |
| | $5.6\times10^{-1}$ | | Fang Lee (2007) | Q | 722 |
| | $4.6\times10^{-2}$ | | Dunnivant et al. (1992) | Q | |
| | $4.3\times10^{-1}$ | | Duchowicz et al. (2020) | ? | 185, 21 |





Table A6.3: Polychlorinated biphenyls (PCBs) (...continued)

| Substance Formula (Trivial Name) [CAS Registry Number] InChIKey | $H_s^{cp}$ (at $T^\ominus$) $\left[\dfrac{\mathrm{mol}}{\mathrm{m^3\,Pa}}\right]$ | $\dfrac{\mathrm{d}\ln H_s^{cp}}{\mathrm{d}(1/T)}$ [K] | Reference | Type | Note |
|---|---|---|---|---|---|
| 2,2',3,3',5,6,6'-heptachlorobiphenyl | $4.1\times10^{-1}$ | | Brunner et al. (1990) | M | |
| $C_{12}H_3Cl_7$ | $4.2\times10^{-3}$ | 7000 | Paasivirta and Sinkkonen (2009) | V | |
| (PCB-179) | $1.1\times10^{-2}$ | | Burkhard et al. (1985) | V | |
| [52663-64-6] | $2.4\times10^{-1}$ | | Keshavarz et al. (2022) | Q | |
| XYHVYEUZLSYHDP-UHFFFAOYSA-N | $1.2\times10^{-1}$ | | Duchowicz et al. (2020) | Q | 299 |
| | $3.1\times10^{-1}$ | | Hilal et al. (2008) | Q | |
| | $3.1\times10^{-1}$ | | Modarresi et al. (2007) | Q | 67 |
| | $2.6\times10^{-1}$ | | Fang Lee (2007) | Q | 721 |
| | $4.8\times10^{-1}$ | | Fang Lee (2007) | Q | 722 |
| | $3.6\times10^{-2}$ | | Dunnivant et al. (1992) | Q | |
| | $4.1\times10^{-1}$ | | Duchowicz et al. (2020) | ? | 185, 21 |
| 2,2',3,4,4',5,5'-heptachlorobiphenyl | $1.7\times10^{-1}$ | 7300 | Li et al. (2003) | L | 366 |
| $C_{12}H_3Cl_7$ | $1.2\times10^{-1}$ | 7900 | Li et al. (2003) | L | 367 |
| (PCB-180) | $2.7\times10^{-2}$ | 17000 | Bamford et al. (2000) | M | |
| [35065-29-3] | $9.9\times10^{-1}$ | | Brunner et al. (1990) | M | |
| WBHQEUPUMONIKF-UHFFFAOYSA-N | $3.1\times10^{-1}$ | | Murphy et al. (1987) | M | 12 |
| | $1.5\times10^{-2}$ | 6900 | Paasivirta and Sinkkonen (2009) | V | |
| | $3.3\times10^{-2}$ | | Burkhard et al. (1985) | V | |
| | $2.5\times10^{-2}$ | 9000 | Paasivirta et al. (1999) | T | |
| | $2.4\times10^{-1}$ | | Keshavarz et al. (2022) | Q | |
| | $2.1\times10^{-1}$ | | Duchowicz et al. (2020) | Q | 299 |
| | $1.4\times10^{-2}$ | | Bhangare et al. (2019) | Q | |
| | $2.1\times10^{-1}$ | | Hilal et al. (2008) | Q | |
| | $3.5\times10^{-1}$ | | Modarresi et al. (2007) | Q | 67 |
| | $2.8\times10^{-1}$ | | Fang Lee (2007) | Q | 721 |
| | $3.0\times10^{-1}$ | | Fang Lee (2007) | Q | 722 |
| | $9.2\times10^{-2}$ | | Dunnivant et al. (1992) | Q | |
| | $9.9\times10^{-1}$ | | Duchowicz et al. (2020) | ? | 185, 21 |
| 2,2',3,4,4',5,6-heptachlorobiphenyl | $1.2\times10^{-2}$ | 7200 | Paasivirta and Sinkkonen (2009) | V | |
| $C_{12}H_3Cl_7$ | $2.4\times10^{-2}$ | | Burkhard et al. (1985) | V | |
| (PCB-181) | $1.6\times10^{-1}$ | | Fang Lee (2007) | Q | 721 |
| [74472-47-2] | $3.8\times10^{-1}$ | | Fang Lee (2007) | Q | 722 |
| DJEUXBQAKBLKPO-UHFFFAOYSA-N | $4.3\times10^{-2}$ | | Dunnivant et al. (1992) | Q | |
| 2,2',3,4,4',5,6'-heptachlorobiphenyl | $1.7\times10^{-2}$ | 12000 | Bamford et al. (2002) | M | |
| $C_{12}H_3Cl_7$ | $1.7\times10^{-2}$ | 7200 | Paasivirta and Sinkkonen (2009) | V | |
| (PCB-182) | $1.0\times10^{-2}$ | | Burkhard et al. (1985) | V | |
| [60145-23-5] | $1.5\times10^{-1}$ | | Fang Lee (2007) | Q | 721 |
| RXRLRYZUMSYVLS-UHFFFAOYSA-N | $2.1\times10^{-1}$ | | Fang Lee (2007) | Q | 722 |
| | $3.8\times10^{-2}$ | | Dunnivant et al. (1992) | Q | |





Table A6.3: Polychlorinated biphenyls (PCBs) (. . . continued)

| Substance Formula (Trivial Name) [CAS Registry Number] InChIKey | $H_s^{cp}$ (at $T^\ominus$) $\left[\dfrac{\mathrm{mol}}{\mathrm{m^3\,Pa}}\right]$ | $\dfrac{\mathrm{d\ln}H_s^{cp}}{\mathrm{d}(1/T)}$ [K] | Reference | Type | Note |
|---|---|---|---|---|---|
| 2,2',3,4,4',5',6-heptachlorobiphenyl | $1.7\times10^{-2}$ | 12000 | Bamford et al. (2002) | M | |
| $C_{12}H_3Cl_7$ | $1.5\times10^{-1}$ | | Murphy et al. (1987) | M | 12 |
| (PCB-183) | $1.6\times10^{-2}$ | | Murphy et al. (1983a) | M | 24 |
| [52663-69-1] | $2.4\times10^{-2}$ | 7400 | Paasivirta and Sinkkonen (2009) | V | |
| KQBFUDNJKCZEDQ-UHFFFAOYSA-N | $2.1\times10^{-2}$ | | Burkhard et al. (1985) | V | |
| | $1.7\times10^{-1}$ | | Fang Lee (2007) | Q | 721 |
| | $2.5\times10^{-1}$ | | Fang Lee (2007) | Q | 722 |
| | $4.9\times10^{-2}$ | | Dunnivant et al. (1992) | Q | |
| 2,2',3,4,4',6,6'-heptachlorobiphenyl | $8.1\times10^{-3}$ | 7200 | Paasivirta and Sinkkonen (2009) | V | |
| $C_{12}H_3Cl_7$ | $7.9\times10^{-3}$ | | Burkhard et al. (1985) | V | |
| (PCB-184) | $9.4\times10^{-2}$ | | Fang Lee (2007) | Q | 721 |
| [74472-48-3] | $1.3\times10^{-1}$ | | Fang Lee (2007) | Q | 722 |
| OBIUJJSQKPGKME-UHFFFAOYSA-N | $2.2\times10^{-2}$ | | Dunnivant et al. (1992) | Q | |
| 2,2',3,4,5,5',6-heptachlorobiphenyl | $6.2\times10^{-1}$ | | Brunner et al. (1990) | M | |
| $C_{12}H_3Cl_7$ | $4.9\times10^{-3}$ | 7000 | Paasivirta and Sinkkonen (2009) | V | |
| (PCB-185) | $2.2\times10^{-2}$ | | Burkhard et al. (1985) | V | |
| [52712-05-7] | $2.4\times10^{-1}$ | | Keshavarz et al. (2022) | Q | |
| PYZHTHZEHQHHEN-UHFFFAOYSA-N | $1.1\times10^{-1}$ | | Duchowicz et al. (2020) | Q | 299 |
| | $1.5\times10^{-1}$ | | Hilal et al. (2008) | Q | |
| | $3.4\times10^{-1}$ | | Modarresi et al. (2007) | Q | 67 |
| | $3.1\times10^{-1}$ | | Fang Lee (2007) | Q | 721 |
| | $4.3\times10^{-1}$ | | Fang Lee (2007) | Q | 722 |
| | $4.6\times10^{-2}$ | | Dunnivant et al. (1992) | Q | |
| | $6.2\times10^{-1}$ | | Duchowicz et al. (2020) | ? | 185, 21 |
| 2,2',3,4,5,6,6'-heptachlorobiphenyl | $9.6\times10^{-4}$ | 6500 | Paasivirta and Sinkkonen (2009) | V | |
| $C_{12}H_3Cl_7$ | $1.3\times10^{-2}$ | | Burkhard et al. (1985) | V | |
| (PCB-186) | $1.7\times10^{-1}$ | | Fang Lee (2007) | Q | 721 |
| [74472-49-4] | $3.7\times10^{-1}$ | | Fang Lee (2007) | Q | 722 |
| FGDPOTMRBQHPJK-UHFFFAOYSA-N | $2.7\times10^{-2}$ | | Dunnivant et al. (1992) | Q | |
| 2,2',3,4',5,5',6-heptachlorobiphenyl | $1.6\times10^{-2}$ | 12000 | Bamford et al. (2000) | M | |
| $C_{12}H_3Cl_7$ | $1.2\times10^{-1}$ | | Murphy et al. (1987) | M | 12 |
| (PCB-187) | $1.3\times10^{-2}$ | 7200 | Paasivirta and Sinkkonen (2009) | V | |
| [52663-68-0] | $2.4\times10^{-2}$ | | Burkhard et al. (1985) | V | |
| UDMZPLROONOSEF-UHFFFAOYSA-N | $2.4\times10^{-1}$ | | Fang Lee (2007) | Q | 721 |
| | $4.3\times10^{-1}$ | | Fang Lee (2007) | Q | 722 |
| | $4.9\times10^{-2}$ | | Dunnivant et al. (1992) | Q | |





Table A6.3: Polychlorinated biphenyls (PCBs) (...continued)

| Substance Formula (Trivial Name) [CAS Registry Number] InChIKey | $H_s^{cp}$ (at $T^{\ominus}$) $\left[\dfrac{\text{mol}}{\text{m}^3\,\text{Pa}}\right]$ | $\dfrac{\text{d}\ln H_s^{cp}}{\text{d}(1/T)}$ [K] | Reference | Type | Note |
|---|---|---|---|---|---|
| 2,2',3,4',5,6,6'-heptachlorobiphenyl | $8.8\times10^{-3}$ | 7500 | Bamford et al. (2000) | M | |
| $C_{12}H_3Cl_7$ | $4.8\times10^{-3}$ | 7100 | Paasivirta and Sinkkonen (2009) | V | |
| (PCB-188) | $8.8\times10^{-3}$ | | Burkhard et al. (1985) | V | |
| [74487-85-7] | $1.6\times10^{-2}$ | | Bhangare et al. (2019) | Q | |
| MMTJWDQKGUNSDK-UHFFFAOYSA-N | $1.3\times10^{-1}$ | | Fang Lee (2007) | Q | 721 |
| | $2.3\times10^{-1}$ | | Fang Lee (2007) | Q | 722 |
| | $2.2\times10^{-2}$ | | Dunnivant et al. (1992) | Q | |
| 2,3,3',4,4',5,5'-heptachlorobiphenyl | $8.4\times10^{-2}$ | | Fang et al. (2006) | M | |
| $C_{12}H_3Cl_7$ | $4.1\times10^{-3}$ | 6300 | Paasivirta and Sinkkonen (2009) | V | |
| (PCB-189) | $1.1\times10^{-1}$ | | Burkhard et al. (1985) | V | |
| [39635-31-9] | $3.4\times10^{-1}$ | | Fang Lee (2007) | Q | 721 |
| XUAWBXBYHDRROL-UHFFFAOYSA-N | $3.0\times10^{-1}$ | | Fang Lee (2007) | Q | 722 |
| | $1.5\times10^{-1}$ | | Dunnivant et al. (1992) | Q | |
| 2,3,3',4,4',5,6-heptachlorobiphenyl | $1.5\times10^{-2}$ | 7000 | Paasivirta and Sinkkonen (2009) | V | |
| $C_{12}H_3Cl_7$ | $9.9\times10^{-2}$ | | Burkhard et al. (1985) | V | |
| (PCB-190) | $1.9\times10^{-1}$ | | Fang Lee (2007) | Q | 721 |
| [41411-64-7] | $5.3\times10^{-1}$ | | Fang Lee (2007) | Q | 722 |
| TYEDCFVCFDKSBK-UHFFFAOYSA-N | $8.8\times10^{-2}$ | | Dunnivant et al. (1992) | Q | |
| 2,3,3',4,4',5',6-heptachlorobiphenyl | $2.1\times10^{-2}$ | 7200 | Paasivirta and Sinkkonen (2009) | V | |
| $C_{12}H_3Cl_7$ | $5.8\times10^{-2}$ | | Burkhard et al. (1985) | V | |
| (PCB-191) | $2.1\times10^{-1}$ | | Fang Lee (2007) | Q | 721 |
| [74472-50-7] | $3.2\times10^{-1}$ | | Fang Lee (2007) | Q | 722 |
| TVFXBXWAXIMLAQ-UHFFFAOYSA-N | $7.4\times10^{-2}$ | | Dunnivant et al. (1992) | Q | |
| 2,3,3',4,5,5',6-heptachlorobiphenyl | $4.9\times10^{-3}$ | 6900 | Paasivirta and Sinkkonen (2009) | V | |
| $C_{12}H_3Cl_7$ | $6.1\times10^{-2}$ | | Burkhard et al. (1985) | V | |
| (PCB-192) | $3.8\times10^{-1}$ | | Fang Lee (2007) | Q | 721 |
| [74472-51-8] | $5.0\times10^{-1}$ | | Fang Lee (2007) | Q | 722 |
| ZUTDUGMNROUBOX-UHFFFAOYSA-N | $5.2\times10^{-2}$ | | Dunnivant et al. (1992) | Q | |
| 2,3,3',4',5,5',6-heptachlorobiphenyl | $3.2\times10^{-2}$ | 17000 | Bamford et al. (2002) | M | |
| $C_{12}H_3Cl_7$ | $7.5\times10^{-3}$ | 6800 | Paasivirta and Sinkkonen (2009) | V | |
| (PCB-193) | $1.0\times10^{-1}$ | | Burkhard et al. (1985) | V | |
| [69782-91-8] | $2.9\times10^{-1}$ | | Fang Lee (2007) | Q | 721 |
| SSTJUBQGYXNFFP-UHFFFAOYSA-N | $5.6\times10^{-1}$ | | Fang Lee (2007) | Q | 722 |
| | $7.3\times10^{-2}$ | | Dunnivant et al. (1992) | Q | |



Table A6.3: Polychlorinated biphenyls (PCBs) (...continued)

| Substance Formula (Trivial Name) [CAS Registry Number] InChIKey | $H_s^{cp}$ (at $T^{\ominus}$) $\left[\dfrac{\text{mol}}{\text{m}^3\,\text{Pa}}\right]$ | $\dfrac{\text{d}\ln H_s^{cp}}{\text{d}(1/T)}$ [K] | Reference | Type | Note |
|---|---|---|---|---|---|
| 2,2',3,3',4,4',5,5'- octachlorobiphenyl | $1.5{\times}10^{-1}$ | 7500 | Li et al. (2003) | L | 366 |
| $C_{12}H_2Cl_8$ | $2.3{\times}10^{-1}$ | 8200 | Li et al. (2003) | L | 367 |
| (PCB-194) | $1.0{\times}10^{-1}$ | 20000 | Bamford et al. (2002) | M | |
| [35694-08-7] | $9.9{\times}10^{-1}$ | | Brunner et al. (1990) | M | |
| DTMRKGRREZAYAP-UHFFFAOYSA-N | $8.0{\times}10^{-3}$ | 6900 | Paasivirta and Sinkkonen (2009) | V | |
| | $2.1{\times}10^{-2}$ | | Burkhard et al. (1985) | V | |
| | $2.4{\times}10^{-1}$ | | Keshavarz et al. (2022) | Q | |
| | $4.0{\times}10^{-1}$ | | Duchowicz et al. (2020) | Q | 184 |
| | $2.3{\times}10^{-1}$ | | Hilal et al. (2008) | Q | |
| | $4.8{\times}10^{-1}$ | | Modarresi et al. (2007) | Q | 67 |
| | $5.6{\times}10^{-1}$ | | Fang Lee (2007) | Q | 721 |
| | $7.1{\times}10^{-1}$ | | Fang Lee (2007) | Q | 722 |
| | | 6500 | Kühne et al. (2005) | Q | |
| | $1.5{\times}10^{-1}$ | | Dunnivant et al. (1992) | Q | |
| | $9.9{\times}10^{-1}$ | | Duchowicz et al. (2020) | ? | 185, 21 |
| | | 6600 | Kühne et al. (2005) | ? | |
| 2,2',3,3',4,4',5,6- octachlorobiphenyl | $7.1{\times}10^{-2}$ | 20000 | Bamford et al. (2000) | M | |
| $C_{12}H_2Cl_8$ | $9.0{\times}10^{-1}$ | | Brunner et al. (1990) | M | |
| (PCB-195) | $7.1{\times}10^{-3}$ | 7100 | Paasivirta and Sinkkonen (2009) | V | |
| [52663-78-2] | $7.8{\times}10^{-2}$ | | Burkhard et al. (1985) | V | |
| JAHJITLFJSDRCG-UHFFFAOYSA-N | $2.4{\times}10^{-1}$ | | Keshavarz et al. (2022) | Q | |
| | $3.1{\times}10^{-1}$ | | Duchowicz et al. (2020) | Q | 299 |
| | $2.4{\times}10^{-1}$ | | Hilal et al. (2008) | Q | |
| | $4.5{\times}10^{-1}$ | | Modarresi et al. (2007) | Q | 67 |
| | $3.1{\times}10^{-1}$ | | Fang Lee (2007) | Q | 721 |
| | $1.0$ | | Fang Lee (2007) | Q | 722 |
| | $8.3{\times}10^{-2}$ | | Dunnivant et al. (1992) | Q | |
| | $9.0{\times}10^{-1}$ | | Duchowicz et al. (2020) | ? | 185, 21 |
| 2,2',3,3',4,4',5,6'- octachlorobiphenyl | $9.9{\times}10^{-1}$ | | Brunner et al. (1990) | M | |
| $C_{12}H_2Cl_8$ | $1.8{\times}10^{-2}$ | 7400 | Paasivirta and Sinkkonen (2009) | V | |
| (PCB-196) | $1.4{\times}10^{-2}$ | | Burkhard et al. (1985) | V | |
| [42740-50-1] | $2.4{\times}10^{-1}$ | | Keshavarz et al. (2022) | Q | |
| BQFCCUSDZLKBJG-UHFFFAOYSA-N | $2.4{\times}10^{-1}$ | | Duchowicz et al. (2020) | Q | |
| | $2.2{\times}10^{-1}$ | | Hilal et al. (2008) | Q | |
| | $5.0{\times}10^{-1}$ | | Modarresi et al. (2007) | Q | 67 |
| | $3.4{\times}10^{-1}$ | | Fang Lee (2007) | Q | 721 |
| | $6.2{\times}10^{-1}$ | | Fang Lee (2007) | Q | 722 |
| | $7.6{\times}10^{-2}$ | | Dunnivant et al. (1992) | Q | |
| | $9.9{\times}10^{-1}$ | | Duchowicz et al. (2020) | ? | 185, 21 |




Table A6.3: Polychlorinated biphenyls (PCBs) (…continued)

| Substance Formula (Trivial Name) [CAS Registry Number] InChIKey | $H_s^{cp}$ (at $T^{\ominus}$) $\left[\dfrac{\mathrm{mol}}{\mathrm{m^3\,Pa}}\right]$ | $\dfrac{\mathrm{d\ln}H_s^{cp}}{\mathrm{d}(1/T)}$ [K] | Reference | Type | Note |
|---|---|---|---|---|---|
| 2,2',3,3',4,4',6,6'-octachlorobiphenyl | $1.3\times10^{-2}$ | 7600 | Paasivirta and Sinkkonen (2009) | V | |
| $C_{12}H_2Cl_8$ | $1.1\times10^{-2}$ | | Burkhard et al. (1985) | V | |
| (PCB-197) | $1.9\times10^{-1}$ | | Fang Lee (2007) | Q | 721 |
| [33091-17-7] | $4.2\times10^{-1}$ | | Fang Lee (2007) | Q | 722 |
| YPDBBDKYNWRFMF-UHFFFAOYSA-N | $3.9\times10^{-2}$ | | Dunnivant et al. (1992) | Q | |
| 2,2',3,3',4,5,5',6-octachlorobiphenyl | $7.0\times10^{-1}$ | | Brunner et al. (1990) | M | |
| $C_{12}H_2Cl_8$ | $3.5\times10^{-3}$ | 7000 | Paasivirta and Sinkkonen (2009) | V | |
| (PCB-198) | $4.8\times10^{-2}$ | | Burkhard et al. (1985) | V | |
| [68194-17-2] | $2.4\times10^{-1}$ | | Keshavarz et al. (2022) | Q | |
| PJHBSPRZHUOIAS-UHFFFAOYSA-N | $1.7\times10^{-1}$ | | Duchowicz et al. (2020) | Q | 299 |
| | $2.5\times10^{-1}$ | | Hilal et al. (2008) | Q | |
| | $4.5\times10^{-1}$ | | Modarresi et al. (2007) | Q | 67 |
| | $6.2\times10^{-1}$ | | Fang Lee (2007) | Q | 721 |
| | $1.0$ | | Fang Lee (2007) | Q | 722 |
| | $6.4\times10^{-2}$ | | Dunnivant et al. (1992) | Q | |
| | $7.0\times10^{-1}$ | | Duchowicz et al. (2020) | ? | 185, 21 |
| 2,2',3,3',4,5,5',6'-octachlorobiphenyl | $9.9\times10^{-1}$ | | Brunner et al. (1990) | M | |
| $C_{12}H_2Cl_8$ | $3.4\times10^{-3}$ | 7000 | Paasivirta and Sinkkonen (2009) | V | |
| (PCB-199) | $2.3\times10^{-2}$ | | Burkhard et al. (1985) | V | |
| [52663-75-9] | $2.4\times10^{-1}$ | | Keshavarz et al. (2022) | Q | |
| HJBYDWKNARZTMJ-UHFFFAOYSA-N | $2.2\times10^{-1}$ | | Duchowicz et al. (2020) | Q | 299 |
| | $2.7\times10^{-1}$ | | Hilal et al. (2008) | Q | |
| | $4.7\times10^{-1}$ | | Modarresi et al. (2007) | Q | 67 |
| | $6.2\times10^{-1}$ | | Fang Lee (2007) | Q | 721 |
| | $9.1\times10^{-1}$ | | Fang Lee (2007) | Q | 722 |
| | $4.3\times10^{-2}$ | | Dunnivant et al. (1992) | Q | |
| | $9.9\times10^{-1}$ | | Duchowicz et al. (2020) | ? | 185, 21 |
| 2,2',3,3',4,5,6,6'-octachlorobiphenyl | $7.6\times10^{-3}$ | 7200 | Paasivirta and Sinkkonen (2009) | V | |
| $C_{12}H_2Cl_8$ | $1.5\times10^{-2}$ | | Burkhard et al. (1985) | V | |
| (PCB-200) | $3.4\times10^{-1}$ | | Fang Lee (2007) | Q | 721 |
| [52663-73-7] | $7.7\times10^{-1}$ | | Fang Lee (2007) | Q | 722 |
| HHXNVASVVVNNDG-UHFFFAOYSA-N | $4.1\times10^{-2}$ | | Dunnivant et al. (1992) | Q | |
| 2,2',3,3',4,5',6,6'-octachlorobiphenyl | $1.0\times10^{-2}$ | 17000 | Bamford et al. (2000) | M | |
| $C_{12}H_2Cl_8$ | $5.8\times10^{-1}$ | | Brunner et al. (1990) | M | |
| (PCB-201) | $1.2\times10^{-2}$ | 7500 | Paasivirta and Sinkkonen (2009) | V | |
| [40186-71-8] | $1.5\times10^{-2}$ | | Burkhard et al. (1985) | V | |
| LJQOBQLZTUSEJA-UHFFFAOYSA-N | $2.4\times10^{-1}$ | | Keshavarz et al. (2022) | Q | |
| | $1.3\times10^{-1}$ | | Duchowicz et al. (2020) | Q | 184 |
| | $1.3\times10^{-2}$ | | Bhangare et al. (2019) | Q | |
| | $2.9\times10^{-1}$ | | Hilal et al. (2008) | Q | |



Table A6.3: Polychlorinated biphenyls (PCBs) (. . . continued)

| Substance Formula (Trivial Name) [CAS Registry Number] InChIKey | $H_s^{cp}$ (at $T^{\ominus}$) $\left[\dfrac{\mathrm{mol}}{\mathrm{m}^3\,\mathrm{Pa}}\right]$ | $\dfrac{\mathrm{d}\ln H_s^{cp}}{\mathrm{d}(1/T)}$ [K] | Reference | Type | Note |
|---|---|---|---|---|---|
| | $5.1\times10^{-1}$ | | Modarresi et al. (2007) | Q | 67 |
| | $3.7\times10^{-1}$ | | Fang Lee (2007) | Q | 721 |
| | $1.1$ | | Fang Lee (2007) | Q | 722 |
| | $7.6\times10^{-2}$ | | Dunnivant et al. (1992) | Q | |
| | $5.8\times10^{-1}$ | | Duchowicz et al. (2020) | ? | 185, 21 |
| 2,2',3,3',5,5',6,6'-octachlorobiphenyl | $5.5\times10^{-1}$ | | Brunner et al. (1990) | M | |
| $C_{12}H_2Cl_8$ | $5.0\times10^{-3}$ | 7300 | Paasivirta and Sinkkonen (2009) | V | |
| (PCB-202) | $2.6\times10^{-2}$ | | Mackay et al. (2006b) | V | |
| [2136-99-4] | $2.6\times10^{-2}$ | | Mackay et al. (1992a) | V | |
| JPOPEORRMSDUIP-UHFFFAOYSA-N | $2.7\times10^{-2}$ | | Shiu and Mackay (1986) | V | |
| | $1.3\times10^{-2}$ | | Burkhard et al. (1985) | V | |
| | $2.4\times10^{-1}$ | | Keshavarz et al. (2022) | Q | |
| | $1.2\times10^{-1}$ | | Duchowicz et al. (2020) | Q | |
| | $3.7\times10^{-1}$ | | Hilal et al. (2008) | Q | |
| | $5.5\times10^{-1}$ | | Modarresi et al. (2007) | Q | 67 |
| | $5.3\times10^{-1}$ | | Fang Lee (2007) | Q | 721 |
| | $1.4$ | | Fang Lee (2007) | Q | 722 |
| | | 4700 | Kühne et al. (2005) | Q | |
| | $4.4\times10^{-2}$ | | Dunnivant et al. (1992) | Q | |
| | $5.5\times10^{-1}$ | | Duchowicz et al. (2020) | ? | 185, 21 |
| | | 5000 | Kühne et al. (2005) | ? | |
| 2,2',3,4,4',5,5',6-octachlorobiphenyl | $3.2\times10^{-2}$ | 7800 | Paasivirta and Sinkkonen (2009) | V | |
| $C_{12}H_2Cl_8$ | $5.0\times10^{-2}$ | | Burkhard et al. (1985) | V | |
| (PCB-203) | $1.3\times10^{-2}$ | | Bhangare et al. (2019) | Q | |
| [52663-76-0] | $3.1\times10^{-1}$ | | Fang Lee (2007) | Q | 721 |
| DCPDZFRGNJDWPP-UHFFFAOYSA-N | $7.7\times10^{-1}$ | | Fang Lee (2007) | Q | 722 |
| | $7.0\times10^{-2}$ | | Dunnivant et al. (1992) | Q | |
| 2,2',3,4,4',5,6,6'-octachlorobiphenyl | $1.1\times10^{-2}$ | 7800 | Paasivirta and Sinkkonen (2009) | V | |
| $C_{12}H_2Cl_8$ | $1.9\times10^{-2}$ | | Burkhard et al. (1985) | V | |
| (PCB-204) | $1.3\times10^{-2}$ | | Bhangare et al. (2019) | Q | |
| [74472-52-9] | $1.7\times10^{-1}$ | | Fang Lee (2007) | Q | 721 |
| JDZUWXRNKHXZFE-UHFFFAOYSA-N | $4.5\times10^{-1}$ | | Fang Lee (2007) | Q | 722 |
| | $2.9\times10^{-2}$ | | Dunnivant et al. (1992) | Q | |
| 2,3,3',4,4',5,5',6-octachlorobiphenyl | $4.4\times10^{-3}$ | 6800 | Paasivirta and Sinkkonen (2009) | V | |
| $C_{12}H_2Cl_8$ | $2.1\times10^{-1}$ | | Burkhard et al. (1985) | V | |
| (PCB-205) | $1.2\times10^{-2}$ | | Bhangare et al. (2019) | Q | |
| [74472-53-0] | $3.8\times10^{-1}$ | | Fang Lee (2007) | Q | 721 |
| VXXBCDUYUQKWCK-UHFFFAOYSA-N | $9.1\times10^{-1}$ | | Fang Lee (2007) | Q | 722 |
| | $1.1\times10^{-1}$ | | Dunnivant et al. (1992) | Q | |



Rolf Sander: Compilation of Henry's law constants 1115

Table A6.3: Polychlorinated biphenyls (PCBs) (...continued)

| Substance<br>Formula<br>(Trivial Name)<br>[CAS Registry Number]<br>InChIKey | $H_s^{cp}$ (at $T^{\ominus}$) $\left[\dfrac{\text{mol}}{\text{m}^3\,\text{Pa}}\right]$ | $\dfrac{\text{d}\ln H_s^{cp}}{\text{d}(1/T)}$ [K] | Reference | Type | Note |
|---|---|---|---|---|---|
| 2,2',3,3',4,4',5,5',6-<br>nonachlorobiphenyl | $2.1\times10^{-3}$ | 7300 | Paasivirta and Sinkkonen (2009) | V | |
| $C_{12}HCl_9$ | 1.2 | | Mackay et al. (2006b) | V | |
| (PCB-206) | $1.2\times10^{-2}$ | | Mackay et al. (1992a) | V | |
| [40186-72-9] | $3.6\times10^{-2}$ | | Burkhard et al. (1985) | V | |
| JFIMDKGRGPNPRQ-UHFFFAOYSA-N | $1.1\times10^{-2}$ | | Bhangare et al. (2019) | Q | |
| | $6.2\times10^{-1}$ | | Fang Lee (2007) | Q | 721 |
| | 2.0 | | Fang Lee (2007) | Q | 722 |
| | $1.1\times10^{-1}$ | | Dunnivant et al. (1992) | Q | |
| 2,2',3,3',4,4',5,6,6'-<br>nonachlorobiphenyl | $1.8\times10^{-3}$ | 7500 | Paasivirta and Sinkkonen (2009) | V | |
| $C_{12}HCl_9$ | $2.8\times10^{-2}$ | | Burkhard et al. (1985) | V | |
| (PCB-207) | $1.1\times10^{-2}$ | | Bhangare et al. (2019) | Q | |
| [52663-79-3] | $3.3\times10^{-1}$ | | Fang Lee (2007) | Q | 721 |
| YGDPIDTZOQGPAX-UHFFFAOYSA-N | 1.4 | | Fang Lee (2007) | Q | 722 |
| | $5.8\times10^{-2}$ | | Dunnivant et al. (1992) | Q | |
| 2,2',3,3',4,5,5',6,6'-<br>nonachlorobiphenyl | $3.0\times10^{-3}$ | 7700 | Paasivirta and Sinkkonen (2009) | V | |
| $C_{12}HCl_9$ | $3.1\times10^{-2}$ | | Burkhard et al. (1985) | V | |
| (PCB-208) | $1.2\times10^{-2}$ | | Bhangare et al. (2019) | Q | |
| [52663-77-1] | $6.7\times10^{-1}$ | | Fang Lee (2007) | Q | 721 |
| XIFFTDRFWYFAPO-UHFFFAOYSA-N | 2.5 | | Fang Lee (2007) | Q | 722 |
| | $5.9\times10^{-2}$ | | Dunnivant et al. (1992) | Q | |
| decachlorobiphenyl | $1.0\times10^{-3}$ | | Duchowicz et al. (2020) | V | 186 |
| $C_{12}Cl_{10}$ | $6.7\times10^{-4}$ | 7200 | Paasivirta and Sinkkonen (2009) | V | |
| (PCB-209) | | | Mackay et al. (2006b) | V | 683 |
| [2051-24-3] | $4.8\times10^{-2}$ | | Mackay et al. (1992a) | V | |
| ONXPZLFXDMAPRO-UHFFFAOYSA-N | $4.8\times10^{-2}$ | | Shiu and Mackay (1986) | V | |
| | $8.0\times10^{-2}$ | | Burkhard et al. (1985) | V | |
| | $2.6\times10^{-1}$ | | Duchowicz et al. (2020) | Q | |
| | $1.0\times10^{-2}$ | | Bhangare et al. (2019) | Q | |
| | $3.1\times10^{-1}$ | | Hilal et al. (2008) | Q | |
| | $6.7\times10^{-1}$ | | Fang Lee (2007) | Q | 721 |
| | 5.0 | | Fang Lee (2007) | Q | 722 |
| | | 6100 | Kühne et al. (2005) | Q | |
| | $8.8\times10^{-2}$ | | Dunnivant et al. (1992) | Q | |
| | | 7300 | Kühne et al. (2005) | ? | |



### A6.4 Oxygenated chlorocarbons (C, H, O, Cl)

Table A6.4: Oxygenated chlorocarbons (C, H, O, Cl)

| Substance<br>Formula<br>(Trivial Name)<br>[CAS Registry Number]<br>InChIKey | $H_s^{cp}$<br>(at $T^{\ominus}$)<br>$\left[\dfrac{\mathrm{mol}}{\mathrm{m^3\,Pa}}\right]$ | $\dfrac{\mathrm{d}\ln H_s^{cp}}{\mathrm{d}(1/T)}$<br><br>[K] | Reference | Type | Note |
|---|---|---|---|---|---|
| phosgene | $5.9\times10^{-4}$ | 3800 | De Bruyn et al. (1995a) | M | |
| $CCl_2O$ | $6.8\times10^{-4}$ | 4200 | Manogue and Pigford (1960) | M | |
| [75-44-5] | $7.0\times10^{-4}$ | | Yaws (2003) | X | 237 |
| YGYAWVDWMABLBF-UHFFFAOYSA-N | $5.3\times10^{-4}$ | | Hayer et al. (2022) | Q | 20 |
| | $8.8\times10^{-3}$ | | Keshavarz et al. (2022) | Q | |
| | $6.1\times10^{-2}$ | | Duchowicz et al. (2020) | Q | |
| | $3.7\times10^{-1}$ | | Wang et al. (2017) | Q | 80, 238 |
| | $1.4\times10^{-2}$ | | Wang et al. (2017) | Q | 80, 239 |
| | $1.1\times10^{-4}$ | | Wang et al. (2017) | Q | 80, 240 |
| | $6.8\times10^{-4}$ | | Gharagheizi et al. (2010) | Q | 246 |
| | $5.9\times10^{-4}$ | | Duchowicz et al. (2020) | ? | 185, 21 |
| | $7.1\times10^{-4}$ | | Yaws (1999) | ? | 21 |
| MCM:CCL3OOH | $2.1\times10^{1}$ | | Wang et al. (2017) | Q | 80, 238 |
| $CHO_2Cl_3$ | 4.9 | | Wang et al. (2017) | Q | 80, 239 |
| DUBMHDTUZCIGCY-UHFFFAOYSA-N | 3.8 | | Wang et al. (2017) | Q | 80, 240 |
| MCM:CH2CLOOH | $1.8\times10^{1}$ | | Wang et al. (2017) | Q | 80, 238 |
| $CH_3O_2Cl$ | $6.2\times10^{1}$ | | Wang et al. (2017) | Q | 80, 239 |
| DUNYOPWYTOSXOC-UHFFFAOYSA-N | $1.8\times10^{1}$ | | Wang et al. (2017) | Q | 80, 240 |
| MCM:CHCL2OOH | $4.7\times10^{1}$ | | Wang et al. (2017) | Q | 80, 238 |
| $CH_2O_2Cl_2$ | $1.3\times10^{2}$ | | Wang et al. (2017) | Q | 80, 239 |
| TUIYYMKEIGYDJB-UHFFFAOYSA-N | 5.6 | | Wang et al. (2017) | Q | 80, 240 |
| MCM:CCL3OH | 4.7 | | Wang et al. (2017) | Q | 80, 238 |
| $CHOCl_3$ | $8.1\times10^{-1}$ | | Wang et al. (2017) | Q | 80, 239 |
| GYLIOGDFGLKMOL-UHFFFAOYSA-N | $4.0\times10^{1}$ | | Wang et al. (2017) | Q | 80, 240 |
| MCM:CH2CLOH | 1.9 | | Wang et al. (2017) | Q | 80, 238 |
| $CH_3OCl$ | $1.9\times10^{1}$ | | Wang et al. (2017) | Q | 80, 239 |
| BCUPGIHTCQJCSI-UHFFFAOYSA-N | $1.1\times10^{2}$ | | Wang et al. (2017) | Q | 80, 240 |
| MCM:CHCL2OH | 6.6 | | Wang et al. (2017) | Q | 80, 238 |
| $CH_2OCl_2$ | $1.2\times10^{1}$ | | Wang et al. (2017) | Q | 80, 239 |
| GJYVZUKSNFSLCL-UHFFFAOYSA-N | $2.8\times10^{1}$ | | Wang et al. (2017) | Q | 80, 240 |
| MCM:CHOCL | $2.2\times10^{-1}$ | | Wang et al. (2017) | Q | 80, 238 |
| CHOCl | $2.0\times10^{-2}$ | | Wang et al. (2017) | Q | 80, 239 |
| GFAUNYMRSKVDJL-UHFFFAOYSA-N | $3.0\times10^{-3}$ | | Wang et al. (2017) | Q | 80, 240 |
| MCM:CLCO2H | $3.2\times10^{1}$ | | Wang et al. (2017) | Q | 80, 238 |
| $CHO_2Cl$ | 8.5 | | Wang et al. (2017) | Q | 80, 239 |
| AOGYCOYQMAVAFD-UHFFFAOYSA-N | $9.1\times10^{1}$ | | Wang et al. (2017) | Q | 80, 240 |



Table A6.4: Oxygenated chlorocarbons (C, H, O, Cl) (...continued)

| Substance<br>Formula<br>(Trivial Name)<br>[CAS Registry Number]<br>InChIKey | $H_s^{cp}$<br>(at $T^\ominus$)<br>$\left[\dfrac{\mathrm{mol}}{\mathrm{m^3\,Pa}}\right]$ | $\dfrac{\mathrm{d}\ln H_s^{cp}}{\mathrm{d}(1/T)}$<br><br>[K] | Reference | Type | Note |
|---|---|---|---|---|---|
| dichloroacetaldehyde<br>$C_2H_2Cl_2O$<br>[79-02-7]<br>NWQWQKUXRJYXFH-UHFFFAOYSA-N | $4.7\times10^{-1}$<br>$7.4\times10^{-1}$<br>$1.1\times10^{-2}$<br>$1.2$ | | Wang et al. (2017)<br>Wang et al. (2017)<br>Wang et al. (2017)<br>HSDB (2015) | Q<br>Q<br>Q<br>Q | 80, 238<br>80, 239<br>80, 240<br>99 |
| chloroacetyl chloride<br>$C_2H_2Cl_2O$<br>[79-04-9]<br>VGCXGMAHQTYDJK-UHFFFAOYSA-N | $2.9\times10^{-1}$<br>$1.3\times10^{-1}$<br>$4.3\times10^{-2}$<br>$4.3\times10^{-2}$ | | Wang et al. (2017)<br>Wang et al. (2017)<br>Wang et al. (2017)<br>HSDB (2015) | Q<br>Q<br>Q<br>Q | 80, 238<br>80, 239<br>80, 240<br>99 |
| chloral hydrate<br>$C_2H_3Cl_3O_2$<br>[302-17-0]<br>RNFNDJAIBTYOQL-UHFFFAOYSA-N | $2.4\times10^{3}$ | | HSDB (2015) | V | |
| chloroacetaldehyde<br>$C_2H_3ClO$<br>[107-20-0]<br>QSKPIOLLBIHNAC-UHFFFAOYSA-N | $1.9\times10^{-1}$<br>$2.3$<br>$3.5$<br>$4.1\times10^{-1}$ | | Wang et al. (2017)<br>Wang et al. (2017)<br>Wang et al. (2017)<br>HSDB (2015) | Q<br>Q<br>Q<br>Q | 80, 238<br>80, 239<br>80, 240<br>99 |
| 2-chloroethanol<br>$C_2H_5ClO$<br>[107-07-3]<br>SZIFAVKTNFCBPC-UHFFFAOYSA-N | $1.3\times10^{1}$<br>$3.3$<br>$1.1$<br>$1.3\times10^{1}$<br>$4.7$<br>$9.5\times10^{1}$<br>$2.8\times10^{1}$<br>$1.4\times10^{1}$<br>$1.7\times10^{1}$ | | Duchowicz et al. (2020)<br>Duchowicz et al. (2020)<br>Wang et al. (2017)<br>Wang et al. (2017)<br>Wang et al. (2017)<br>HSDB (2015)<br>Modarresi et al. (2007)<br>Yaffe et al. (2003)<br>Katritzky et al. (1998) | V<br>Q<br>Q<br>Q<br>Q<br>Q<br>Q<br>Q<br>Q | 186<br><br>80, 238<br>80, 239<br>80, 240<br>99<br>67<br>248, 249<br> |
| 2,2-dichloroethanol<br>$C_2H_4Cl_2O$<br>[598-38-9]<br>IDJOCJAIQSKSOP-UHFFFAOYSA-N | $3.7\times10^{-1}$<br>$3.7\times10^{-1}$<br>$3.7\times10^{-1}$<br>$3.2$<br>$2.6\times10^{1}$<br>$6.9$ | | Burkholder et al. (2019)<br>Burkholder et al. (2015)<br>O'Farrell and Waghorne (2010)<br>Wang et al. (2017)<br>Wang et al. (2017)<br>Wang et al. (2017) | L<br>L<br>M<br>Q<br>Q<br>Q | <br><br><br>80, 238<br>80, 239<br>80, 240 |
| 2-chloroethanol-d4<br>$ClC_2D_4OH$<br>[117067-62-6]<br>SZIFAVKTNFCBPC-LNLMKGTHSA-N | $5.0$ | $8700$ | Hiatt (2013) | M | |
| MCM:CCL2CL2OOH<br>$C_2H_2O_2Cl_4$<br>SWNDCAXLXSSRFN-UHFFFAOYSA-N | $3.9\times10^{2}$<br>$7.1\times10^{1}$<br>$1.6\times10^{1}$ | | Wang et al. (2017)<br>Wang et al. (2017)<br>Wang et al. (2017) | Q<br>Q<br>Q | 80, 238<br>80, 239<br>80, 240 |
| MCM:CCL3CH2OOH<br>$C_2H_3O_2Cl_3$<br>RJIYMCLTECJZCN-UHFFFAOYSA-N | $1.7\times10^{1}$<br>$1.8\times10^{1}$<br>$2.6\times10^{1}$ | | Wang et al. (2017)<br>Wang et al. (2017)<br>Wang et al. (2017) | Q<br>Q<br>Q | 80, 238<br>80, 239<br>80, 240 |



Table A6.4: Oxygenated chlorocarbons (C, H, O, Cl) (...continued)

| Substance Formula (Trivial Name) [CAS Registry Number] InChIKey | $H_s^{cp}$ (at $T^{\ominus}$) $\left[\dfrac{\mathrm{mol}}{\mathrm{m^3\,Pa}}\right]$ | $\dfrac{\mathrm{d}\ln H_s^{cp}}{\mathrm{d}(1/T)}$ [K] | Reference | Type | Note |
|---|---|---|---|---|---|
| MCM:CCL3CO3H | $2.6\times10^2$ | | Wang et al. (2017) | Q | 80, 238 |
| $C_2HO_3Cl_3$ | 5.9 | | Wang et al. (2017) | Q | 80, 239 |
| DRTNVNKYVPOWCY-UHFFFAOYSA-N | 1.4 | | Wang et al. (2017) | Q | 80, 240 |
| MCM:CH2CL3OOH | $1.2\times10^2$ | | Wang et al. (2017) | Q | 80, 238 |
| $C_2H_3O_2Cl_3$ | $3.8\times10^1$ | | Wang et al. (2017) | Q | 80, 239 |
| CHVIFDTZRLKJET-UHFFFAOYSA-N | $1.3\times10^1$ | | Wang et al. (2017) | Q | 80, 240 |
| MCM:CH2CLCOOH | $1.6\times10^1$ | | Wang et al. (2017) | Q | 80, 238 |
| $C_2H_5O_2Cl$ | $6.3\times10^1$ | | Wang et al. (2017) | Q | 80, 239 |
| PIXGNJBYAUTOCI-UHFFFAOYSA-N | $5.1\times10^1$ | | Wang et al. (2017) | Q | 80, 240 |
| MCM:CH3CCL2OOH | $1.2\times10^1$ | | Wang et al. (2017) | Q | 80, 238 |
| $C_2H_4O_2Cl_2$ | $1.0\times10^1$ | | Wang et al. (2017) | Q | 80, 239 |
| YUVMNUKOYYKKNS-UHFFFAOYSA-N | 8.9 | | Wang et al. (2017) | Q | 80, 240 |
| MCM:CH3CHCLOOH | $1.9\times10^1$ | | Wang et al. (2017) | Q | 80, 238 |
| $C_2H_5O_2Cl$ | $2.0\times10^1$ | | Wang et al. (2017) | Q | 80, 239 |
| MCYQTMMYWIXRIL-UHFFFAOYSA-N | 5.6 | | Wang et al. (2017) | Q | 80, 240 |
| MCM:CHCL2CO3H | $5.1\times10^2$ | | Wang et al. (2017) | Q | 80, 238 |
| $C_2H_2O_3Cl_2$ | $4.3\times10^1$ | | Wang et al. (2017) | Q | 80, 239 |
| VWNSYHZUZUFFDM-UHFFFAOYSA-N | 8.9 | | Wang et al. (2017) | Q | 80, 240 |
| MCM:CHCL2COOH | $3.9\times10^1$ | | Wang et al. (2017) | Q | 80, 238 |
| $C_2H_4O_2Cl_2$ | $1.1\times10^2$ | | Wang et al. (2017) | Q | 80, 239 |
| GJBKLBBJOKITMY-UHFFFAOYSA-N | $2.7\times10^1$ | | Wang et al. (2017) | Q | 80, 240 |
| MCM:CHCL3OOH | $4.3\times10^2$ | | Wang et al. (2017) | Q | 80, 238 |
| $C_2H_3O_2Cl_3$ | $1.2\times10^2$ | | Wang et al. (2017) | Q | 80, 239 |
| AMASFZACKVPSTN-UHFFFAOYSA-N | $4.1\times10^1$ | | Wang et al. (2017) | Q | 80, 240 |
| MCM:CLETO3H | $2.1\times10^2$ | | Wang et al. (2017) | Q | 80, 238 |
| $C_2H_3O_3Cl$ | $5.5\times10^1$ | | Wang et al. (2017) | Q | 80, 239 |
| JSQPGNJFQUOEPA-UHFFFAOYSA-N | $1.4\times10^1$ | | Wang et al. (2017) | Q | 80, 240 |
| MCM:DICLETO2H | $1.3\times10^2$ | | Wang et al. (2017) | Q | 80, 238 |
| $C_2H_4O_2Cl_2$ | $7.3\times10^1$ | | Wang et al. (2017) | Q | 80, 239 |
| UPXDMONURWDHKL-UHFFFAOYSA-N | $4.8\times10^1$ | | Wang et al. (2017) | Q | 80, 240 |
| MCM:C2CL2OH2 | $1.4\times10^5$ | | Wang et al. (2017) | Q | 80, 238 |
| $C_2H_4O_2Cl_2$ | $1.3\times10^4$ | | Wang et al. (2017) | Q | 80, 239 |
| HHQQTVXFKGIICR-UHFFFAOYSA-N | $1.2\times10^2$ | | Wang et al. (2017) | Q | 80, 240 |
| MCM:C2CL2OHOOH | $2.8\times10^6$ | | Wang et al. (2017) | Q | 80, 238 |
| $C_2H_4O_3Cl_2$ | $1.7\times10^4$ | | Wang et al. (2017) | Q | 80, 239 |
| NCBKAYIJWMIXBU-UHFFFAOYSA-N | $1.4\times10^4$ | | Wang et al. (2017) | Q | 80, 240 |
| MCM:C2CL32OH | $1.9\times10^5$ | | Wang et al. (2017) | Q | 80, 238 |
| $C_2H_3O_2Cl_3$ | $5.1\times10^3$ | | Wang et al. (2017) | Q | 80, 239 |
| MYDJEUINZIFHKK-UHFFFAOYSA-N | $2.8\times10^7$ | | Wang et al. (2017) | Q | 80, 240 |



Table A6.4: Oxygenated chlorocarbons (C, H, O, Cl) (. . . continued)

| Substance Formula (Trivial Name) [CAS Registry Number] InChIKey | $H_s^{cp}$ (at $T^{\ominus}$) $\left[\dfrac{\text{mol}}{\text{m}^3\,\text{Pa}}\right]$ | $\dfrac{\mathrm{d}\ln H_s^{cp}}{\mathrm{d}(1/T)}$ [K] | Reference | Type | Note |
|---|---|---|---|---|---|
| MCM:C2CL3HOOOH | $3.1\times10^6$ | | Wang et al. (2017) | Q | 80, 238 |
| $C_2H_3O_3Cl_3$ | $1.0\times10^4$ | | Wang et al. (2017) | Q | 80, 239 |
| KHLAQOPQQHOMKQ-UHFFFAOYSA-N | $4.3\times10^6$ | | Wang et al. (2017) | Q | 80, 240 |
| MCM:C2CL3OHOOH | $2.6\times10^6$ | | Wang et al. (2017) | Q | 80, 238 |
| $C_2H_3O_3Cl_3$ | $1.1\times10^4$ | | Wang et al. (2017) | Q | 80, 239 |
| CEKNFZNLLTXCLL-UHFFFAOYSA-N | $1.1\times10^4$ | | Wang et al. (2017) | Q | 80, 240 |
| trichloroethanol | 1.7 | | Wang et al. (2017) | Q | 80, 238 |
| $C_2H_3OCl_3$ | 3.9 | | Wang et al. (2017) | Q | 80, 239 |
| [115-20-8] | 3.2 | | Wang et al. (2017) | Q | 80, 240 |
| KPWDGTGXUYRARH-UHFFFAOYSA-N | | | | | |
| MCM:CCLOHCOOH | $1.9\times10^5$ | | Wang et al. (2017) | Q | 80, 238 |
| $C_2H_5O_3Cl$ | $9.6\times10^3$ | | Wang et al. (2017) | Q | 80, 239 |
| MIFGQALJBIDZDW-UHFFFAOYSA-N | $1.0\times10^5$ | | Wang et al. (2017) | Q | 80, 240 |
| MCM:CH2CL3OH | $2.9\times10^1$ | | Wang et al. (2017) | Q | 80, 238 |
| $C_2H_3OCl_3$ | $1.3\times10^1$ | | Wang et al. (2017) | Q | 80, 239 |
| HYCHPIPDVAXCCJ-UHFFFAOYSA-N | $1.5\times10^2$ | | Wang et al. (2017) | Q | 80, 240 |
| MCM:CH2OHCCLOH | $4.2\times10^3$ | | Wang et al. (2017) | Q | 80, 238 |
| $C_2H_5O_2Cl$ | $4.2\times10^3$ | | Wang et al. (2017) | Q | 80, 239 |
| VVKFXODVPQSIHU-UHFFFAOYSA-N | $9.3\times10^2$ | | Wang et al. (2017) | Q | 80, 240 |
| MCM:CH3CCL2OH | 1.6 | | Wang et al. (2017) | Q | 80, 238 |
| $C_2H_4OCl_2$ | 2.4 | | Wang et al. (2017) | Q | 80, 239 |
| DNBABSPIEDTPHK-UHFFFAOYSA-N | 8.1 | | Wang et al. (2017) | Q | 80, 240 |
| MCM:CH3CHCLOH | 1.8 | | Wang et al. (2017) | Q | 80, 238 |
| $C_2H_5OCl$ | 8.1 | | Wang et al. (2017) | Q | 80, 239 |
| KJESGYZFVCIMDE-UHFFFAOYSA-N | $4.6\times10^1$ | | Wang et al. (2017) | Q | 80, 240 |
| MCM:CHCL2CL2OH | $1.1\times10^2$ | | Wang et al. (2017) | Q | 80, 238 |
| $C_2H_2OCl_4$ | $2.2\times10^1$ | | Wang et al. (2017) | Q | 80, 239 |
| LQINPQOSBLVJBS-UHFFFAOYSA-N | $3.7\times10^2$ | | Wang et al. (2017) | Q | 80, 240 |
| MCM:CHCL3OH | $9.1\times10^1$ | | Wang et al. (2017) | Q | 80, 238 |
| $C_2H_3OCl_3$ | $5.6\times10^1$ | | Wang et al. (2017) | Q | 80, 239 |
| NIBKDWIGIKUFKL-UHFFFAOYSA-N | $3.4\times10^1$ | | Wang et al. (2017) | Q | 80, 240 |
| MCM:CL2OHCH2OH | $4.5\times10^3$ | | Wang et al. (2017) | Q | 80, 238 |
| $C_2H_4O_2Cl_2$ | $1.6\times10^3$ | | Wang et al. (2017) | Q | 80, 239 |
| ZOZPDNLNQWCVSW-UHFFFAOYSA-N | $1.6\times10^4$ | | Wang et al. (2017) | Q | 80, 240 |
| MCM:CL2OHCO3H | $9.3\times10^4$ | | Wang et al. (2017) | Q | 80, 238 |
| $C_2H_2O_4Cl_2$ | 4.7 | | Wang et al. (2017) | Q | 80, 239 |
| MACSTFWYVGNISK-UHFFFAOYSA-N | 5.9 | | Wang et al. (2017) | Q | 80, 240 |
| MCM:CL2OHOOH | $1.7\times10^5$ | | Wang et al. (2017) | Q | 80, 238 |
| $C_2H_4O_3Cl_2$ | $4.0\times10^3$ | | Wang et al. (2017) | Q | 80, 239 |
| WOJAYLCSWSBEJV-UHFFFAOYSA-N | $4.1\times10^5$ | | Wang et al. (2017) | Q | 80, 240 |



Table A6.4: Oxygenated chlorocarbons (C, H, O, Cl) (. . . continued)

| Substance Formula (Trivial Name) [CAS Registry Number] InChIKey | $H_s^{cp}$ (at $T^{\ominus}$) $\left[\dfrac{\text{mol}}{\text{m}^3\,\text{Pa}}\right]$ | $\dfrac{\text{d}\ln H_s^{cp}}{\text{d}(1/T)}$ [K] | Reference | Type | Note |
|---|---|---|---|---|---|
| MCM:CLOHCO3H | $1.0 \times 10^5$ | | Wang et al. (2017) | Q | 80, 238 |
| $C_2H_3O_4Cl$ | $2.5 \times 10^2$ | | Wang et al. (2017) | Q | 80, 239 |
| IODLIDNNOHQSDW-UHFFFAOYSA-N | $1.3 \times 10^3$ | | Wang et al. (2017) | Q | 80, 240 |
| MCM:COHCCLOOH | $1.2 \times 10^5$ | | Wang et al. (2017) | Q | 80, 238 |
| $C_2H_5O_3Cl$ | $5.0 \times 10^3$ | | Wang et al. (2017) | Q | 80, 239 |
| WXPZQCJQULOKCY-UHFFFAOYSA-N | $4.4 \times 10^3$ | | Wang et al. (2017) | Q | 80, 240 |
| MCM:DICLETOH | $2.6 \times 10^1$ | | Wang et al. (2017) | Q | 80, 238 |
| $C_2H_4OCl_2$ | $3.8 \times 10^1$ | | Wang et al. (2017) | Q | 80, 239 |
| QXBDFCZHAAOUBY-UHFFFAOYSA-N | $3.1 \times 10^1$ | | Wang et al. (2017) | Q | 80, 240 |
| MCM:TCE2OH | $2.9 \times 10^5$ | | Wang et al. (2017) | Q | 80, 238 |
| $C_2H_2O_2Cl_4$ | $2.0 \times 10^3$ | | Wang et al. (2017) | Q | 80, 239 |
| BVBMRJKFECUARX-UHFFFAOYSA-N | $4.2 \times 10^6$ | | Wang et al. (2017) | Q | 80, 240 |
| MCM:TCEOHOOH | $3.4 \times 10^6$ | | Wang et al. (2017) | Q | 80, 238 |
| $C_2H_2O_3Cl_4$ | $4.3 \times 10^3$ | | Wang et al. (2017) | Q | 80, 239 |
| YKIXQGIQZGTABG-UHFFFAOYSA-N | $1.4 \times 10^6$ | | Wang et al. (2017) | Q | 80, 240 |
| MCM:CCLOHCHO | $8.3 \times 10^1$ | | Wang et al. (2017) | Q | 80, 238 |
| $C_2H_3O_2Cl$ | $8.9 \times 10^1$ | | Wang et al. (2017) | Q | 80, 239 |
| RYRGLFUQPWFXSD-UHFFFAOYSA-N | $4.4$ | | Wang et al. (2017) | Q | 80, 240 |
| MCM:CL2OHCHO | $7.6 \times 10^1$ | | Wang et al. (2017) | Q | 80, 238 |
| $C_2H_2O_2Cl_2$ | $1.0$ | | Wang et al. (2017) | Q | 80, 239 |
| LCOUNIIIIWZOM-UHFFFAOYSA-N | $6.0 \times 10^{-1}$ | | Wang et al. (2017) | Q | 80, 240 |
| MCM:CH3COCL | $1.5 \times 10^{-1}$ | | Wang et al. (2017) | Q | 80, 238 |
| $C_2H_3OCl$ | $2.6 \times 10^{-2}$ | | Wang et al. (2017) | Q | 80, 239 |
| WETWJCDKMRHUPV-UHFFFAOYSA-N | $8.7 \times 10^{-3}$ | | Wang et al. (2017) | Q | 80, 240 |
| MCM:CHCL2COCL | $6.8 \times 10^{-1}$ | | Wang et al. (2017) | Q | 80, 238 |
| $C_2HOCl_3$ | $6.6 \times 10^{-2}$ | | Wang et al. (2017) | Q | 80, 239 |
| FBCCMZVIWNDFMO-UHFFFAOYSA-N | $3.6 \times 10^{-3}$ | | Wang et al. (2017) | Q | 80, 240 |
| MCM:CLCOCH2OOH | $1.1 \times 10^4$ | | Wang et al. (2017) | Q | 80, 238 |
| $C_2H_3O_3Cl$ | $2.1 \times 10^2$ | | Wang et al. (2017) | Q | 80, 239 |
| ARKKUYXZQNWAAF-UHFFFAOYSA-N | $8.0$ | | Wang et al. (2017) | Q | 80, 240 |
| MCM:CLCOCL2OOH | $1.6 \times 10^4$ | | Wang et al. (2017) | Q | 80, 238 |
| $C_2HO_3Cl_3$ | $3.1 \times 10^1$ | | Wang et al. (2017) | Q | 80, 239 |
| FXUYMCPKRPYDCU-UHFFFAOYSA-N | $2.1$ | | Wang et al. (2017) | Q | 80, 240 |
| MCM:CLCOCLOOH | $2.5 \times 10^4$ | | Wang et al. (2017) | Q | 80, 238 |
| $C_2H_2O_3Cl_2$ | $3.9 \times 10^2$ | | Wang et al. (2017) | Q | 80, 239 |
| WROSSTYSMNZXTC-UHFFFAOYSA-N | $2.6 \times 10^1$ | | Wang et al. (2017) | Q | 80, 240 |
| MCM:CLGLYOX | $2.2 \times 10^2$ | | Wang et al. (2017) | Q | 80, 238 |
| $C_2O_2Cl_2$ | $1.3 \times 10^{-1}$ | | Wang et al. (2017) | Q | 80, 239 |
| CTSLXHKWHWQRSH-UHFFFAOYSA-N | $3.6 \times 10^{-4}$ | | Wang et al. (2017) | Q | 80, 240 |





Table A6.4: Oxygenated chlorocarbons (C, H, O, Cl) (...continued)

| Substance<br>Formula<br>(Trivial Name)<br>[CAS Registry Number]<br>InChIKey | $H_s^{cp}$<br>(at $T^\ominus$)<br>$\left[\dfrac{\text{mol}}{\text{m}^3\,\text{Pa}}\right]$ | $\dfrac{\text{d}\ln H_s^{cp}}{\text{d}(1/T)}$<br><br>[K] | Reference | Type | Note |
|---|---|---|---|---|---|
| MCM:CCL2OHCOCL<br>$C_2HO_2Cl_3$<br>WBBOIYQGQVKNEO-UHFFFAOYSA-N | $1.3\times10^2$<br>1.2<br>$1.7\times10^{-1}$ | | Wang et al. (2017)<br>Wang et al. (2017)<br>Wang et al. (2017) | Q<br>Q<br>Q | 80, 238<br>80, 239<br>80, 240 |
| MCM:CCLOHCOCL<br>$C_2H_2O_2Cl_2$<br>NABBVDLPPITPRF-UHFFFAOYSA-N | $1.4\times10^2$<br>$5.6\times10^1$<br>3.5 | | Wang et al. (2017)<br>Wang et al. (2017)<br>Wang et al. (2017) | Q<br>Q<br>Q | 80, 238<br>80, 239<br>80, 240 |
| MCM:CH2OHCOCL<br>$C_2H_3O_2Cl$<br>LCIMJULVQOQTEZ-UHFFFAOYSA-N | $2.1\times10^1$<br>$1.1\times10^2$<br>$4.2\times10^1$ | | Wang et al. (2017)<br>Wang et al. (2017)<br>Wang et al. (2017) | Q<br>Q<br>Q | 80, 238<br>80, 239<br>80, 240 |
| MCM:CL1GLYOX<br>$C_2HO_2Cl$<br>ZNBGBHISQKMEPA-UHFFFAOYSA-N | $1.4\times10^2$<br>1.5<br>$4.2\times10^{-3}$ | | Wang et al. (2017)<br>Wang et al. (2017)<br>Wang et al. (2017) | Q<br>Q<br>Q | 80, 238<br>80, 239<br>80, 240 |
| MCM:CLOHCO2H<br>$C_2H_3O_3Cl$<br>SORBEIKZIPTJRS-UHFFFAOYSA-N | $8.5\times10^4$<br>$3.6\times10^4$<br>$1.4\times10^5$ | | Wang et al. (2017)<br>Wang et al. (2017)<br>Wang et al. (2017) | Q<br>Q<br>Q | 80, 238<br>80, 239<br>80, 240 |
| 1,1,1-trichloro-2-propanone<br>$C_3H_3Cl_3O$<br>[918-00-3]<br>SMZHKGXSEAGRTI-UHFFFAOYSA-N | 4.5 | | HSDB (2015) | Q | 99 |
| 1,1-dichloro-2-propanone<br>$C_3H_4Cl_2O$<br>(1,1-dichloroacetone)<br>[513-88-2]<br>CSVFWMMPUJDVKH-UHFFFAOYSA-N | $3.3\times10^{-1}$<br>$3.3\times10^{-1}$<br>$3.3\times10^{-1}$<br>1.6 | | Burkholder et al. (2019)<br>Burkholder et al. (2015)<br>O'Farrell and Waghorne (2010)<br>HSDB (2015) | L<br>L<br>M<br>Q | <br><br><br>99 |
| carbonochloridic acid,<br>2-chloroethyl ester<br>$C_3H_4Cl_2O_2$<br>(chloroethyl chloroformate)<br>[627-11-2]<br>SVDDJQGVOFZBNX-UHFFFAOYSA-N | $9.0\times10^{-3}$ | | HSDB (2015) | Q | 99 |
| carbonochloridic acid, ethyl ester<br>$C_3H_5ClO_2$<br>(ethyl chloroformate)<br>[541-41-3]<br>RIFGWPKJUGCATF-UHFFFAOYSA-N | $3.2\times10^{-3}$ | | HSDB (2015) | Q | 99 |
| 2-chloropropanoic acid<br>$C_3H_5ClO_2$<br>[598-78-7]<br>GAWAYYRQGQZKCR-UHFFFAOYSA-N | $3.8\times10^1$ | | HSDB (2015) | Q | 99 |



Table A6.4: Oxygenated chlorocarbons (C, H, O, Cl) (...continued)

| Substance<br>Formula<br>(Trivial Name)<br>[CAS Registry Number]<br>InChIKey | $H_s^{cp}$<br>(at $T^\ominus$)<br>$\left[\dfrac{\mathrm{mol}}{\mathrm{m^3\,Pa}}\right]$ | $\dfrac{\mathrm{d}\ln H_s^{cp}}{\mathrm{d}(1/T)}$<br><br>[K] | Reference | Type | Note |
|---|---|---|---|---|---|
| 2,3-dichloro-1-propanol<br>$C_3H_6Cl_2O$<br>[616-23-9]<br>ZXCYIJGIGSDJQQ-UHFFFAOYSA-N | 7.8<br>$6.9\times10^1$<br>$1.3\times10^1$<br>$3.3\times10^3$ | | Wang et al. (2017)<br>Wang et al. (2017)<br>Wang et al. (2017)<br>HSDB (2015) | Q<br>Q<br>Q<br>Q | 80, 238<br>80, 239<br>80, 240<br>99 |
| 1,3-dichloro-2-propanol<br>$C_3H_6Cl_2O$<br>[96-23-1]<br>DEWLEGDTCGBNGU-UHFFFAOYSA-N | 5.8<br>$1.6\times10^1$<br>$2.0\times10^1$<br>$4.9\times10^1$<br>$1.6\times10^1$<br>$2.6\times10^1$<br>$3.0\times10^1$<br>$1.7\times10^1$ | | Meylan and Howard (1991)<br>HSDB (2015)<br>Raventos-Duran et al. (2010)<br>Raventos-Duran et al. (2010)<br>Raventos-Duran et al. (2010)<br>Hilal et al. (2008)<br>Modarresi et al. (2007)<br>Meylan and Howard (1991) | V<br>Q<br>Q<br>Q<br>Q<br>Q<br>Q<br>Q | <br>99<br>271, 243<br>244<br>245<br><br>67<br> |
| 3-chloro-1,2-propanediol<br>$C_3H_7ClO_2$<br>[96-24-2]<br>SSZWWUDQMAHNAQ-UHFFFAOYSA-N | $1.6\times10^2$ | | HSDB (2015) | Q | 99 |
| 1-chloro-2-propanol<br>$C_3H_7ClO$<br>[127-00-4]<br>YYTSGNJTASLUOY-UHFFFAOYSA-N | 5.8 | | HSDB (2015) | Q | 99 |
| 2-chloro-1-propanol<br>$C_3H_7ClO$<br>[78-89-7]<br>VZIQXGLTRZLBEX-UHFFFAOYSA-N | 5.8 | | HSDB (2015) | Q | 99 |
| 3-chloro-1-propanol<br>$C_3H_7ClO$<br>[627-30-5]<br>LAMUXTNQCICZQX-UHFFFAOYSA-N | $1.3\times10^1$ | | Ebert et al. (2023) | ? | 316 |
| trichloroethanal<br>$CCl_3CHO$<br>(trichloroacetaldehyde; chloral)<br>[75-87-6]<br>HFFLGKNGCAIQMO-UHFFFAOYSA-N | $3.4\times10^3$<br>$3.4\times10^3$<br>$3.4\times10^3$<br>$2.2\times10^3$<br>$1.3\times10^{-1}$<br>$2.1\times10^{-1}$<br>$1.6\times10^{-2}$<br>$3.2\times10^{-3}$<br>$1.2\times10^3$<br>$3.1\times10^2$<br>3.1<br><br>$1.7\times10^3$<br>$3.4\times10^3$ | 3500<br>3500<br>3500<br><br><br><br><br><br><br><br><br>1700<br><br><br>3500 | Burkholder et al. (2019)<br>Burkholder et al. (2015)<br>Betterton and Hoffmann (1988)<br>Keshavarz et al. (2022)<br>Duchowicz et al. (2020)<br>Wang et al. (2017)<br>Wang et al. (2017)<br>Wang et al. (2017)<br>Raventos-Duran et al. (2010)<br>Raventos-Duran et al. (2010)<br>Raventos-Duran et al. (2010)<br>Kühne et al. (2005)<br>Meylan and Howard (1991)<br>Duchowicz et al. (2020)<br>Kühne et al. (2005) | L<br>L<br>M<br>Q<br>Q<br>Q<br>Q<br>Q<br>Q<br>Q<br>Q<br>Q<br>Q<br>?<br>? | 460<br>460<br>460<br><br><br>80, 238<br>80, 239<br>80, 240<br>242, 243<br>244<br>245<br><br><br>185, 21<br> |





Table A6.4: Oxygenated chlorocarbons (C, H, O, Cl) (. . . continued)

| Substance<br>Formula<br>(Trivial Name)<br>[CAS Registry Number]<br>InChIKey | $H_s^{cp}$<br>(at $T^\ominus$)<br>$\left[\dfrac{\mathrm{mol}}{\mathrm{m^3\,Pa}}\right]$ | $\dfrac{\mathrm{d}\ln H_s^{cp}}{\mathrm{d}(1/T)}$<br><br>[K] | Reference | Type | Note |
|---|---|---|---|---|---|
| chloro-2-propanone | $5.8\times10^{-1}$ | 5400 | Burkholder et al. (2019) | L | |
| CH$_2$ClCOCH$_3$ | $5.8\times10^{-1}$ | 5400 | Burkholder et al. (2015) | L | |
| (chloroacetone) | $5.8\times10^{-1}$ | 5400 | Sander et al. (2011) | L | |
| [78-95-5] | $5.8\times10^{-1}$ | 5400 | Betterton (1991) | M | |
| BULLHNJGPPOUOX-UHFFFAOYSA-N | $1.2\times10^{-1}$ | | Keshavarz et al. (2022) | Q | |
| | $8.8\times10^{-2}$ | | Duchowicz et al. (2020) | Q | 299 |
| | $1.3\times10^{-1}$ | | Wang et al. (2017) | Q | 80, 238 |
| | 2.0 | | Wang et al. (2017) | Q | 80, 239 |
| | 1.2 | | Wang et al. (2017) | Q | 80, 240 |
| | 1.6 | | Raventos-Duran et al. (2010) | Q | 271, 243 |
| | 1.2 | | Raventos-Duran et al. (2010) | Q | 244 |
| | $6.2\times10^{-1}$ | | Raventos-Duran et al. (2010) | Q | 245 |
| | $8.8\times10^{-1}$ | | Hilal et al. (2008) | Q | |
| | $6.8\times10^{-1}$ | | Modarresi et al. (2007) | Q | 67 |
| | | 4400 | Kühne et al. (2005) | Q | |
| | $6.0\times10^{-1}$ | | Duchowicz et al. (2020) | ? | 185, 21 |
| | | 5500 | Kühne et al. (2005) | ? | |
| chloroethanoic acid | $1.1\times10^{3}$ | 9700 | Burkholder et al. (2019) | L | |
| CH$_2$ClCOOH | $1.1\times10^{3}$ | 9700 | Burkholder et al. (2015) | L | |
| (chloroacetic acid) | $1.1\times10^{3}$ | 9700 | Sander et al. (2011) | L | |
| [79-11-8] | $1.1\times10^{3}$ | 9700 | Bowden et al. (1998a) | M | |
| FOCAUTSVDIKZOP-UHFFFAOYSA-N | $1.1\times10^{3}$ | | Keshavarz et al. (2022) | Q | |
| | $1.3\times10^{2}$ | | Duchowicz et al. (2020) | Q | |
| | $2.0\times10^{2}$ | | Wang et al. (2017) | Q | 80, 238 |
| | $3.6\times10^{3}$ | | Wang et al. (2017) | Q | 80, 239 |
| | $1.1\times10^{3}$ | | Wang et al. (2017) | Q | 80, 240 |
| | $1.2\times10^{3}$ | | Raventos-Duran et al. (2010) | Q | 271, 243 |
| | $1.6\times10^{3}$ | | Raventos-Duran et al. (2010) | Q | 244 |
| | $4.9\times10^{1}$ | | Raventos-Duran et al. (2010) | Q | 245 |
| | $8.8\times10^{2}$ | | Hilal et al. (2008) | Q | |
| | $3.1\times10^{2}$ | | Modarresi et al. (2007) | Q | 67 |
| | | 8100 | Kühne et al. (2005) | Q | |
| | $1.1\times10^{3}$ | | Duchowicz et al. (2020) | ? | 185, 21 |
| | | 9400 | Kühne et al. (2005) | ? | |
| dichloroethanoic acid | $1.2\times10^{3}$ | 8000 | Burkholder et al. (2019) | L | |
| CHCl$_2$COOH | $1.2\times10^{3}$ | 8000 | Burkholder et al. (2015) | L | |
| (dichloroacetic acid) | $1.2\times10^{3}$ | 8000 | Sander et al. (2011) | L | |
| [79-43-6] | $1.2\times10^{3}$ | 8000 | Bowden et al. (1998a) | M | |
| JXTHNDFMNIQAHM-UHFFFAOYSA-N | $5.6\times10^{2}$ | | Keshavarz et al. (2022) | Q | |
| | $3.9\times10^{2}$ | | Duchowicz et al. (2020) | Q | 299 |
| | $6.6\times10^{2}$ | | Wang et al. (2017) | Q | 80, 238 |
| | $1.4\times10^{3}$ | | Wang et al. (2017) | Q | 80, 239 |
| | $3.8\times10^{2}$ | | Wang et al. (2017) | Q | 80, 240 |
| | $2.5\times10^{3}$ | | Raventos-Duran et al. (2010) | Q | 271, 243 |
| | $6.2\times10^{2}$ | | Raventos-Duran et al. (2010) | Q | 244 |
| | $1.6\times10^{2}$ | | Raventos-Duran et al. (2010) | Q | 245 |





Table A6.4: Oxygenated chlorocarbons (C, H, O, Cl) (. . . continued)

| Substance Formula (Trivial Name) [CAS Registry Number] InChIKey | $H_s^{cp}$ (at $T^\ominus$) $\left[\dfrac{\text{mol}}{\text{m}^3\,\text{Pa}}\right]$ | $\dfrac{\text{d}\ln H_s^{cp}}{\text{d}(1/T)}$ [K] | Reference | Type | Note |
|---|---|---|---|---|---|
| | $3.9 \times 10^2$ | | Hilal et al. (2008) | Q | |
| | $8.5 \times 10^2$ | | Modarresi et al. (2007) | Q | 67 |
| | | 8400 | Kühne et al. (2005) | Q | |
| | $2.6 \times 10^1$ | | Katritzky et al. (1998) | Q | |
| | $1.2 \times 10^3$ | | Duchowicz et al. (2020) | ? | 185, 21 |
| | | 8000 | Kühne et al. (2005) | ? | |
| trichloroethanoic acid CCl$_3$COOH (trichloroacetic acid) [76-03-9] YNJBWRMUSHSURL-UHFFFAOYSA-N | $7.3 \times 10^2$ | 8700 | Burkholder et al. (2019) | L | |
| | $7.3 \times 10^2$ | 8700 | Burkholder et al. (2015) | L | |
| | $7.3 \times 10^2$ | 8700 | Sander et al. (2011) | L | |
| | $7.3 \times 10^2$ | 8700 | Bowden et al. (1998b) | M | |
| | $5.6 \times 10^2$ | | Keshavarz et al. (2022) | Q | |
| | $2.3 \times 10^2$ | | Duchowicz et al. (2020) | Q | |
| | $9.9 \times 10^2$ | | Raventos-Duran et al. (2010) | Q | 271, 243 |
| | 3.9 | | Raventos-Duran et al. (2010) | Q | 244 |
| | $3.9 \times 10^2$ | | Raventos-Duran et al. (2010) | Q | 245 |
| | 4.7 | | Hilal et al. (2008) | Q | |
| | $7.6 \times 10^2$ | | Modarresi et al. (2007) | Q | 67 |
| | | 8800 | Kühne et al. (2005) | Q | |
| | $7.3 \times 10^2$ | | Duchowicz et al. (2020) | ? | 185, 21 |
| | | 8600 | Kühne et al. (2005) | ? | |
| 2,2-dichloro-propanoic acid C$_3$H$_4$Cl$_2$O$_2$ [75-99-0] NDUPDOJHUQKPAG-UHFFFAOYSA-N | $1.7 \times 10^2$ | | Duchowicz et al. (2020) | V | 186 |
| | $3.5 \times 10^8$ | | Mackay et al. (2006d) | V | |
| | $1.8 \times 10^2$ | | Duchowicz et al. (2020) | Q | |
| trichloroacetylchloride CCl$_3$COCl [76-02-8] PVFOMCVHYWHZJE-UHFFFAOYSA-N | $2.0 \times 10^{-2}$ | | Mirabel et al. (1996) | M | |
| | $2.0 \times 10^{-2}$ | | De Bruyn et al. (1995a) | M | 449 |
| | $2.0 \times 10^{-2}$ | | George et al. (1994a) | M | |
| | $1.2 \times 10^{-2}$ | | Keshavarz et al. (2022) | Q | |
| | $7.3 \times 10^{-2}$ | | Duchowicz et al. (2020) | Q | 184 |
| | $3.4 \times 10^{-1}$ | | HSDB (2015) | Q | 99 |
| | $1.9 \times 10^{-2}$ | | Duchowicz et al. (2020) | ? | 185, 21 |
| hexachloroacetone C$_3$Cl$_6$O [116-16-5] DOJXGHGHTWFZHK-UHFFFAOYSA-N | $1.0 \times 10^2$ | | Zhang et al. (2010) | Q | 287, 288 |
| | $9.0 \times 10^{-4}$ | | Zhang et al. (2010) | Q | 287, 289 |
| | $6.2 \times 10^{-2}$ | | Zhang et al. (2010) | Q | 287, 290 |
| | $1.8 \times 10^{-1}$ | | Zhang et al. (2010) | Q | 287, 291 |
| (chloromethyl)-oxirane C$_3$H$_5$ClO (epichlorohydrin) [106-89-8] BRLQWZUYTZBJKN-UHFFFAOYSA-N | $2.9 \times 10^{-1}$ | | Welke et al. (1998) | M | |
| | $3.2 \times 10^{-1}$ | | Duchowicz et al. (2020) | V | 186 |
| | $3.3 \times 10^{-1}$ | | HSDB (2015) | V | |
| | $3.0 \times 10^{-1}$ | | Mackay et al. (2006c) | V | |
| | $3.0 \times 10^{-1}$ | | Mackay et al. (1993) | V | |
| | $2.8 \times 10^{-1}$ | | Goldstein (1982) | X | 446 |
| | $2.8 \times 10^{-1}$ | 3700 | Goldstein (1982) | X | 298 |
| | $1.3 \times 10^{-1}$ | | Duchowicz et al. (2020) | Q | |
| | $9.9 \times 10^{-2}$ | | Hilal et al. (2008) | Q | |
| | $1.7 \times 10^{-1}$ | | Modarresi et al. (2007) | Q | 67 |



Table A6.4: Oxygenated chlorocarbons (C, H, O, Cl) (. . . continued)

| Substance Formula (Trivial Name) [CAS Registry Number] InChIKey | $H_s^{cp}$ (at $T^\ominus$) $\left[\dfrac{\text{mol}}{\text{m}^3\,\text{Pa}}\right]$ | $\dfrac{\text{d}\ln H_s^{cp}}{\text{d}(1/T)}$ [K] | Reference | Type | Note |
|---|---|---|---|---|---|
| methyl chloroethanoate $C_3H_5ClO_2$ [96-34-4] QABLOFMHHSOFRJ-UHFFFAOYSA-N | $4.9\times10^{-1}$ $4.2\times10^{-1}$ $4.1\times10^{-2}$ $3.3\times10^{-1}$ $2.3\times10^{-1}$ $1.2\times10^{-1}$ | 5600 | Brockbank (2013) Duchowicz et al. (2020) HSDB (2015) Duchowicz et al. (2020) Hilal et al. (2008) Modarresi et al. (2007) | L V V Q Q Q | 1 186 |
| MCM:CL12CO3H $C_3H_4O_3Cl_2$ VMJCNPRTZBGZGF-UHFFFAOYSA-N | $1.6\times10^3$ $1.1\times10^2$ 2.3 | | Wang et al. (2017) Wang et al. (2017) Wang et al. (2017) | Q Q Q | 80, 238 80, 239 80, 240 |
| MCM:CL12PRAOOH $C_3H_6O_2Cl_2$ WGVINOMLJVZPKZ-UHFFFAOYSA-N | $1.0\times10^2$ $2.5\times10^2$ $1.5\times10^2$ | | Wang et al. (2017) Wang et al. (2017) Wang et al. (2017) | Q Q Q | 80, 238 80, 239 80, 240 |
| MCM:CL12PRBOOH $C_3H_6O_2Cl_2$ OMIMXWULHYQJIC-UHFFFAOYSA-N | $7.1\times10^1$ $1.8\times10^1$ 9.3 | | Wang et al. (2017) Wang et al. (2017) Wang et al. (2017) | Q Q Q | 80, 238 80, 239 80, 240 |
| MCM:CL12PRCOOH $C_3H_6O_2Cl_2$ ULSSYZYIMOSSQB-UHFFFAOYSA-N | $1.2\times10^2$ $3.8\times10^1$ $3.8\times10^1$ | | Wang et al. (2017) Wang et al. (2017) Wang et al. (2017) | Q Q Q | 80, 238 80, 239 80, 240 |
| MCM:CL12PRBOH $C_3H_6OCl_2$ QEZDCTNHTRSNMD-UHFFFAOYSA-N | $1.4\times10^1$ $1.5\times10^1$ $2.6\times10^1$ | | Wang et al. (2017) Wang et al. (2017) Wang et al. (2017) | Q Q Q | 80, 238 80, 239 80, 240 |
| MCM:CL12PRCOH $C_3H_6OCl_2$ FLTSEOGWHPJWRV-UHFFFAOYSA-N | $2.3\times10^1$ $1.8\times10^1$ $4.9\times10^1$ | | Wang et al. (2017) Wang et al. (2017) Wang et al. (2017) | Q Q Q | 80, 238 80, 239 80, 240 |
| MCM:CL12PRCHO $C_3H_4OCl_2$ IZRKUJREXIKAQM-UHFFFAOYSA-N | 1.3 3.5 $5.5\times10^{-1}$ | | Wang et al. (2017) Wang et al. (2017) Wang et al. (2017) | Q Q Q | 80, 238 80, 239 80, 240 |
| MCM:CH3CLCOCL $C_3H_4OCl_2$ JEQDSBVHLKBEIZ-UHFFFAOYSA-N | $2.7\times10^{-1}$ $3.8\times10^{-2}$ $1.1\times10^{-2}$ | | Wang et al. (2017) Wang et al. (2017) Wang et al. (2017) | Q Q Q | 80, 238 80, 239 80, 240 |
| MCM:CH3COCLOOH $C_3H_5O_3Cl$ HLLYIZVQRAGDFF-UHFFFAOYSA-N | $1.0\times10^4$ $3.5\times10^2$ $1.0\times10^2$ | | Wang et al. (2017) Wang et al. (2017) Wang et al. (2017) | Q Q Q | 80, 238 80, 239 80, 240 |
| MCM:CLCOCLMOOH $C_3H_4O_3Cl_2$ RUPGTCZALLJKQM-UHFFFAOYSA-N | $1.4\times10^4$ $6.0\times10^1$ 1.3 | | Wang et al. (2017) Wang et al. (2017) Wang et al. (2017) | Q Q Q | 80, 238 80, 239 80, 240 |
| MCM:CL12CO2H $C_3H_4O_2Cl_2$ GKFWNPPZHDYVLI-UHFFFAOYSA-N | $1.3\times10^3$ $3.5\times10^3$ $8.1\times10^2$ | | Wang et al. (2017) Wang et al. (2017) Wang et al. (2017) | Q Q Q | 80, 238 80, 239 80, 240 |



Table A6.4: Oxygenated chlorocarbons (C, H, O, Cl) (...continued)

| Substance Formula (Trivial Name) [CAS Registry Number] InChIKey | $H_s^{cp}$ (at $T^{\ominus}$) $\left[\dfrac{\text{mol}}{\text{m}^3\,\text{Pa}}\right]$ | $\dfrac{\text{d}\ln H_s^{cp}}{\text{d}(1/T)}$ [K] | Reference | Type | Note |
|---|---|---|---|---|---|
| 3-chloro-2-butanone C$_4$H$_7$ClO [4091-39-8] OIMRLHCSLQUXLL-UHFFFAOYSA-N | $9.5\times10^{-2}$ | | Ebert et al. (2023) | ? | 316 |
| ethyl chloroethanoate C$_4$H$_7$ClO$_2$ [105-39-5] VEUUMBGHMNQHGO-UHFFFAOYSA-N | $2.6\times10^{-1}$ $2.4\times10^{-1}$ $1.5\times10^{-1}$ $1.2\times10^{-1}$ $1.1\times10^{-1}$ $1.1\times10^{-1}$ $2.4\times10^{-1}$ | 6100 | Brockbank (2013) Duchowicz et al. (2020) Duchowicz et al. (2020) HSDB (2015) Hilal et al. (2008) Modarresi et al. (2007) Bartelt-Hunt et al. (2008) | L V Q Q Q Q ? | 1 186 99 67 21 |
| carbonochloridic acid, 1-methylethyl ester C$_4$H$_7$ClO$_2$ [108-23-6] IVRIRQXJSNCSPQ-UHFFFAOYSA-N | $2.4\times10^{-3}$ | | HSDB (2015) | Q | 99 |
| chloroacetic acid anhydride C$_4$H$_4$Cl$_2$O$_3$ [541-88-8] PNVPNXKRAUBJGW-UHFFFAOYSA-N | 2.2 | | HSDB (2015) | Q | 99 |
| chlorobutanol C$_4$H$_9$ClO [1320-66-7] OSASVXMJTNOKOY-UHFFFAOYSA-N | 4.5 | | HSDB (2015) | Q | 99 |
| methyl 2-chloroacetoacetate C$_5$H$_7$ClO$_3$ [4755-81-1] GYQRIAVRKLRQKP-UHFFFAOYSA-N | 2.3 | | Ebert et al. (2023) | ? | 318 |
| 3-chloro-4-(dichloromethyl)-2-(5H)-furanone C$_5$H$_3$Cl$_3$O$_2$ [122551-89-7] WNQKLIFDPFSPIZ-UHFFFAOYSA-N | 1.5 | | HSDB (2015) | Q | 99 |
| 3-chloro-4-(dichloromethyl)-5-hydroxy-2-(5H)-furanone C$_5$H$_3$Cl$_3$O$_3$ [77439-76-0] WNTRMRXAGJOLCU-UHFFFAOYSA-N | $3.9\times10^4$ | | HSDB (2015) | Q | 99 |
| 1,2,4-trichloro-2-methyl-3-pentanone C$_6$H$_9$Cl$_3$O [145556-04-3] KAUWEMQJFDRELR-UHFFFAOYSA-N | 1.9 1.9 $1.1\times10^2$ $4.1\times10^{-1}$ | | Zhang et al. (2010) Zhang et al. (2010) Zhang et al. (2010) Zhang et al. (2010) | Q Q Q Q | 287, 288 287, 289 287, 290 287, 291 |



Table A6.4: Oxygenated chlorocarbons (C, H, O, Cl) (... continued)

| Substance Formula (Trivial Name) [CAS Registry Number] InChIKey | $H_s^{cp}$ (at $T^{\ominus}$) $\left[\dfrac{\text{mol}}{\text{m}^3\,\text{Pa}}\right]$ | $\dfrac{\text{d}\ln H_s^{cp}}{\text{d}(1/T)}$ [K] | Reference | Type | Note |
|---|---|---|---|---|---|
| bis(2-chloroisopropyl) ether | $3.0\times10^{-2}$ | | Zhang et al. (2010) | Q | 287, 288 |
| $C_6H_{12}Cl_2O$ | $3.3\times10^{-4}$ | | Zhang et al. (2010) | Q | 287, 289 |
| [39638-32-9] | $9.5\times10^{-1}$ | | Zhang et al. (2010) | Q | 287, 290 |
| BULHJTXRZFEUDQ-UHFFFAOYSA-N | $1.6\times10^{-3}$ | | Zhang et al. (2010) | Q | 287, 291 |
| butyl 2,2,3,4,4-pentachloro-3-butenoate | $3.9\times10^{-1}$ | | Zhang et al. (2010) | Q | 287, 288 |
| $C_8H_9Cl_5O_2$ | $3.9\times10^{-2}$ | | Zhang et al. (2010) | Q | 287, 289 |
| [75147-20-5] | $7.9\times10^{-2}$ | | Zhang et al. (2010) | Q | 287, 290 |
| JZJILZTVTMMGAR-UHFFFAOYSA-N | $1.8\times10^{-1}$ | | Zhang et al. (2010) | Q | 287, 291 |
| 3-(2,2-dichlorovinyl)-2,2-dimethylcyclopropane carbonyl chloride | $1.6\times10^{-2}$ | | Zhang et al. (2010) | Q | 287, 288 |
| $C_8H_9Cl_3O$ | $9.7\times10^{-3}$ | | Zhang et al. (2010) | Q | 287, 289 |
| [52314-67-7] | $1.6\times10^{-1}$ | | Zhang et al. (2010) | Q | 287, 290 |
| CHLAOFANYRDCPD-UHFFFAOYSA-N | $8.8\times10^{-1}$ | | Zhang et al. (2010) | Q | 287, 291 |
| 3-(2,2-dichlorovinyl)-2,2-dimethylcyclopropane carboxylic acid | $1.9\times10^{1}$ | | Zhang et al. (2010) | Q | 287, 288 |
| $C_8H_{10}Cl_2O_2$ | $9.0\times10^{1}$ | | Zhang et al. (2010) | Q | 287, 289 |
| [55701-05-8] | $6.1\times10^{4}$ | | Zhang et al. (2010) | Q | 287, 290 |
| LLMLSUSAKZVFOA-UHFFFAOYSA-N | $6.1\times10^{1}$ | | Zhang et al. (2010) | Q | 287, 291 |
| hexanoic acid, 3,3-dimethyl-4,6,6,6-tetrachloro, methyl ester | $6.7\times10^{-1}$ | | Zhang et al. (2010) | Q | 287, 288 |
| $C_9H_{14}Cl_4O_2$ | $2.7\times10^{-1}$ | | Zhang et al. (2010) | Q | 287, 289 |
| [64667-33-0] | $6.1\times10^{2}$ | | Zhang et al. (2010) | Q | 287, 290 |
| POFHGKISWXYKLB-UHFFFAOYSA-N | $1.8\times10^{-2}$ | | Zhang et al. (2010) | Q | 287, 291 |
| methyl 3-(2,2-dichlorovinyl)-2,2-dimethylcyclopropanecarboxylate | $6.1\times10^{-2}$ | | Zhang et al. (2010) | Q | 287, 288 |
| $C_9H_{12}Cl_2O_2$ | $5.2\times10^{-2}$ | | Zhang et al. (2010) | Q | 287, 289 |
| [61898-95-1] | $1.3\times10^{-1}$ | | Zhang et al. (2010) | Q | 287, 290 |
| QJOOIMSFFIUFKX-UHFFFAOYSA-N | $1.7\times10^{-1}$ | | Zhang et al. (2010) | Q | 287, 291 |
| oxychlordane | $6.0\times10^{-2}$ | 4300 | Paasivirta et al. (1999) | T | |
| $C_{10}H_4Cl_8O$ | $1.1\times10^{2}$ | | HSDB (2015) | Q | 99 |
| [27304-13-8] | | | | | |
| VWGNQYSIWFHEQU-UHFFFAOYSA-N | | | | | |
| kepone | $1.8\times10^{2}$ | | Duchowicz et al. (2020) | V | 186 |
| $C_{10}Cl_{10}O$ | $1.8\times10^{2}$ | | HSDB (2015) | V | |
| [143-50-0] | $2.0\times10^{2}$ | | Mackay et al. (2006d) | V | |
| LHHGDZSESBACKH-UHFFFAOYSA-N | $1.5\times10^{2}$ | | Duchowicz et al. (2020) | Q | |




Table A6.4: Oxygenated chlorocarbons (C, H, O, Cl) (. . . continued)

| Substance Formula (Trivial Name) [CAS Registry Number] InChIKey | $H_s^{cp}$ (at $T^\ominus$) $\left[\dfrac{\text{mol}}{\text{m}^3\,\text{Pa}}\right]$ | $\dfrac{\text{d}\ln H_s^{cp}}{\text{d}(1/T)}$ [K] | Reference | Type | Note |
|---|---|---|---|---|---|
| bis-(chloromethyl) ether | $2.3\times10^{-3}$ | | Duchowicz et al. (2020) | V | 186 |
| $C_2H_4Cl_2O$ | $4.8\times10^{-2}$ | | Mackay et al. (2006c) | V | |
| [542-88-1] | $4.8\times10^{-2}$ | | Mackay et al. (1993) | V | |
| HRQGCQVOJVTVLU-UHFFFAOYSA-N | $4.7\times10^{-3}$ | | Ryan et al. (1988) | C | |
| | $1.9\times10^{-2}$ | | Duchowicz et al. (2020) | Q | |
| | $4.3\times10^{-2}$ | | Modarresi et al. (2007) | Q | 67 |
| 1,5-dichloro-3-oxapentane | $4.8\times10^{-1}$ | 6200 | Brockbank (2013) | L | 1 |
| $C_4H_8Cl_2O$ | $5.8\times10^{-1}$ | | Duchowicz et al. (2020) | V | 186 |
| (bis-(2-chloroethyl)-ether) | $3.4\times10^{-1}$ | | HSDB (2015) | V | |
| [111-44-4] | $3.5\times10^{-1}$ | | Mackay et al. (2006c) | V | |
| ZNSMNVMLTJELDZ-UHFFFAOYSA-N | $3.4\times10^{-2}$ | | Lide and Frederikse (1995) | V | |
| | $3.5\times10^{-1}$ | | Mackay et al. (1993) | V | |
| | $4.6\times10^{-1}$ | | Goldstein (1982) | X | 446 |
| | $4.7\times10^{-1}$ | 4100 | Goldstein (1982) | X | 298 |
| | $3.7\times10^{-1}$ | | Harrison et al. (1993) | C | |
| | 8.6 | | Ryan et al. (1988) | C | |
| | $3.4\times10^{-2}$ | | Duchowicz et al. (2020) | Q | |
| | $5.2\times10^{-2}$ | | Zhang et al. (2010) | Q | 287, 288 |
| | $2.8\times10^{-1}$ | | Zhang et al. (2010) | Q | 287, 289 |
| | $4.4\times10^{-2}$ | | Zhang et al. (2010) | Q | 287, 290 |
| | $4.6\times10^{-3}$ | | Zhang et al. (2010) | Q | 287, 291 |
| | $2.9\times10^{-1}$ | | Hilal et al. (2008) | Q | |
| | $3.7\times10^{-1}$ | | Modarresi et al. (2007) | Q | 67 |
| | | 6000 | Kühne et al. (2005) | Q | |
| | | 6000 | Kühne et al. (2005) | ? | |
| 2-chloro-1,1-dimethoxyethane | $3.2\times10^{-1}$ | | Ebert et al. (2023) | ? | 318 |
| $C_4H_9ClO_2$ | | | | | |
| [97-97-2] | | | | | |
| CRZJPEIBPQWDGJ-UHFFFAOYSA-N | | | | | |
| (2-chloroethoxy)-ethene | $1.1\times10^{-3}$ | | Duchowicz et al. (2020) | V | 186 |
| $C_4H_7ClO$ | $1.1\times10^{-3}$ | | HSDB (2015) | V | |
| (2-chloroethylvinylether) | $3.9\times10^{-2}$ | | Mackay et al. (2006c) | V | |
| [110-75-8] | $3.9\times10^{-2}$ | | Mackay et al. (1993) | V | |
| DNJRKFKAFWSXSE-UHFFFAOYSA-N | $1.1\times10^{-3}$ | | Goldstein (1982) | X | 446 |
| | $3.1\times10^{-2}$ | 2500 | Goldstein (1982) | X | 298 |
| | $4.0\times10^{-2}$ | | Ryan et al. (1988) | C | |
| | $4.9\times10^{-2}$ | | Duchowicz et al. (2020) | Q | |
| | $2.3\times10^{-3}$ | | Hilal et al. (2008) | Q | |
| | $2.1\times10^{-2}$ | | Modarresi et al. (2007) | Q | 67 |





Table A6.4: Oxygenated chlorocarbons (C, H, O, Cl) (...continued)

| Substance<br>Formula<br>(Trivial Name)<br>[CAS Registry Number]<br>InChIKey | $H_s^{cp}$ (at $T^{\ominus}$) $\left[\dfrac{\mathrm{mol}}{\mathrm{m^3\,Pa}}\right]$ | $\dfrac{\mathrm{d}\ln H_s^{cp}}{\mathrm{d}(1/T)}$ [K] | Reference | Type | Note |
|---|---|---|---|---|---|
| bis-(2-chloroethoxy)-methane | 2.6 | | Duchowicz et al. (2020) | V | 186 |
| C$_5$H$_{10}$Cl$_2$O$_2$ | 2.5 | | HSDB (2015) | V | |
| [111-91-1] | 2.2×10$^1$ | | Mackay et al. (2006c) | V | |
| NLXGURFLBLRZRO-UHFFFAOYSA-N | 2.2 | | Mackay et al. (1993) | V | |
| | 8.8 | | Goldstein (1982) | X | 446 |
| | 2.6×10$^1$ | 5500 | Goldstein (1982) | X | 298 |
| | 3.7×10$^1$ | | Ryan et al. (1988) | C | |
| | 2.9×10$^{-1}$ | | Duchowicz et al. (2020) | Q | |
| | 3.4 | | Hilal et al. (2008) | Q | |
| bis-(2-chloroisopropyl) ether | 4.2×10$^{-1}$ | | Kawamoto and Urano (1989) | M | |
| C$_6$H$_{12}$Cl$_2$O | 1.3×10$^{-1}$ | | HSDB (2015) | V | |
| (DCIP) | 9.6×10$^{-2}$ | | Mackay et al. (2006c) | V | |
| [108-60-1] | 9.6×10$^{-2}$ | | Mackay et al. (1993) | V | |
| QCFYJCYNJLBDRT-UHFFFAOYSA-N | 6.5×10$^{-2}$ | | Goldstein (1982) | X | 446 |
| | 6.4×10$^{-2}$ | 2800 | Goldstein (1982) | X | 298 |
| | 8.6×10$^{-3}$ | | Ryan et al. (1988) | C | |
| | 7.2×10$^{-2}$ | | Hilal et al. (2008) | Q | |
| | 2.9×10$^{-1}$ | | Modarresi et al. (2007) | Q | 67 |
| 1,2-bis(2-chloroethoxy)ethane | 1.3×10$^1$ | | HSDB (2015) | V | |
| C$_6$H$_{12}$Cl$_2$O$_2$ | 4.7 | | Modarresi et al. (2007) | Q | 67 |
| [112-26-5] | | | | | |
| AGYUOJIYYGGHKV-UHFFFAOYSA-N | | | | | |
| 2-hydroxychlorobenzene | 1.5 | | Sheikheldin et al. (2001) | M | 12 |
| C$_6$H$_5$ClO | 3.6 | 5700 | Tabai et al. (1997) | M | 11 |
| (o-chlorophenol) | 1.5 | | Mackay et al. (2006c) | V | |
| [95-57-8] | 1.2 | | Fogg and Sangster (2003) | V | 729 |
| ISPYQTSUDJAMAB-UHFFFAOYSA-N | 1.8×10$^1$ | | Lide and Frederikse (1995) | V | |
| | 1.5 | | Mackay et al. (1995) | V | |
| | 1.5 | | Shiu et al. (1994) | V | |
| | 8.8×10$^{-1}$ | | Abraham et al. (1994a) | R | |
| | 1.2 | | Goldstein (1982) | X | 446 |
| | 1.2 | 4600 | Goldstein (1982) | X | 298 |
| | 1.8×10$^1$ | | Howard (1989) | X | 418 |
| | 2.1 | | Ryan et al. (1988) | C | |
| | 2.2 | | Keshavarz et al. (2022) | Q | |
| | 8.2 | | Duchowicz et al. (2020) | Q | 299 |
| | 4.2 | | Hilal et al. (2008) | Q | |
| | 4.6 | | Modarresi et al. (2007) | Q | 67 |
| | | 6200 | Kühne et al. (2005) | Q | |
| | 9.2×10$^{-1}$ | | Yaffe et al. (2003) | Q | 248, 249 |
| | 1.8×10$^2$ | | Nirmalakhandan et al. (1997) | Q | |
| | 8.8×10$^{-1}$ | | Duchowicz et al. (2020) | ? | 185, 21 |
| | 8.8×10$^{-1}$ | | HSDB (2015) | ? | 419 |
| | | 5600 | Kühne et al. (2005) | ? | |
| | 2.6×10$^{-1}$ | | Yaws (1999) | ? | 21 |
| | 1.0 | | Chiou et al. (1980) | ? | 79 |





Table A6.4: Oxygenated chlorocarbons (C, H, O, Cl) (...continued)

| Substance Formula (Trivial Name) [CAS Registry Number] InChIKey | $H_s^{cp}$ (at $T^{\ominus}$) $\left[\dfrac{\text{mol}}{\text{m}^3\,\text{Pa}}\right]$ | $\dfrac{\text{d}\ln H_s^{cp}}{\text{d}(1/T)}$ [K] | Reference | Type | Note |
|---|---|---|---|---|---|
| 3-hydroxychlorobenzene | 7.2 | | Chao et al. (2017) | M | |
| $C_6H_5ClO$ | $3.4\times10^1$ | 6400 | Tabai et al. (1997) | M | 11 |
| ($m$-chlorophenol) | 4.9 | | Mackay et al. (2006c) | V | |
| [108-43-0] | 7.3 | | Fogg and Sangster (2003) | V | |
| HORNXRXVQWOLPJ-UHFFFAOYSA-N | $1.8\times10^1$ | | Lide and Frederikse (1995) | V | |
| | 4.9 | | Mackay et al. (1995) | V | |
| | 4.9 | | Shiu et al. (1994) | V | |
| | $2.9\times10^1$ | | Abraham et al. (1994a) | R | |
| | $1.8\times10^1$ | | Howard (1989) | X | 418 |
| | 4.3 | | Keshavarz et al. (2022) | Q | |
| | $9.9\times10^1$ | | Duchowicz et al. (2020) | Q | 299 |
| | $1.6\times10^1$ | | Hilal et al. (2008) | Q | |
| | $2.5\times10^1$ | | Modarresi et al. (2007) | Q | 67 |
| | | 6200 | Kühne et al. (2005) | Q | |
| | $3.0\times10^1$ | | Yaffe et al. (2003) | Q | 248, 249 |
| | $1.3\times10^1$ | | English and Carroll (2001) | Q | 230, 231 |
| | $2.4\times10^1$ | | Katritzky et al. (1998) | Q | |
| | $1.8\times10^2$ | | Nirmalakhandan et al. (1997) | Q | |
| | $2.9\times10^1$ | | Duchowicz et al. (2020) | ? | 185, 21 |
| | $2.9\times10^1$ | | HSDB (2015) | ? | 419 |
| | | 6100 | Kühne et al. (2005) | ? | |
| | 4.8 | | Yaws (1999) | ? | 21, 12 |
| 4-hydroxychlorobenzene | 6.7 | | Chao et al. (2017) | M | |
| $C_6H_5ClO$ | $1.4\times10^3$ | 11000 | Tabai et al. (1997) | M | 11 |
| ($p$-chlorophenol) | $1.6\times10^1$ | | HSDB (2015) | V | |
| [106-48-9] | $1.1\times10^1$ | | Mackay et al. (2006c) | V | |
| WXNZTHHGJRFXKQ-UHFFFAOYSA-N | $1.2\times10^1$ | | Fogg and Sangster (2003) | V | |
| | $1.8\times10^1$ | | Lide and Frederikse (1995) | V | |
| | $1.1\times10^1$ | | Mackay et al. (1995) | V | |
| | $1.1\times10^1$ | | Shiu et al. (1994) | V | |
| | $5.8\times10^1$ | | Abraham et al. (1994a) | R | |
| | $1.8\times10^1$ | | Howard (1989) | X | 418 |
| | 4.3 | | Keshavarz et al. (2022) | Q | |
| | $1.5\times10^2$ | | Duchowicz et al. (2020) | Q | 184 |
| | $2.4\times10^1$ | | Li et al. (2014) | Q | 241 |
| | $1.3\times10^1$ | | Hilal et al. (2008) | Q | |
| | 9.8 | | Modarresi et al. (2007) | Q | 67 |
| | | 6200 | Kühne et al. (2005) | Q | |
| | $5.8\times10^1$ | | Yaffe et al. (2003) | Q | 248, 249 |
| | $3.4\times10^1$ | | Yao et al. (2002) | Q | 229 |
| | $2.3\times10^1$ | | English and Carroll (2001) | Q | 230, 260 |
| | $3.0\times10^1$ | | Katritzky et al. (1998) | Q | |
| | $1.8\times10^2$ | | Nirmalakhandan et al. (1997) | Q | |
| | $1.6\times10^1$ | | Duchowicz et al. (2020) | ? | 185, 21 |
| | | 6400 | Kühne et al. (2005) | ? | |
| | 7.8 | | Yaws (1999) | ? | 21, 12 |





Table A6.4: Oxygenated chlorocarbons (C, H, O, Cl) (...continued)

| Substance Formula (Trivial Name) [CAS Registry Number] InChIKey | $H_s^{cp}$ (at $T^{\ominus}$) $\left[\dfrac{\text{mol}}{\text{m}^3\,\text{Pa}}\right]$ | $\dfrac{\text{d}\ln H_s^{cp}}{\text{d}(1/T)}$ [K] | Reference | Type | Note |
|---|---|---|---|---|---|
| | $1.1\times10^1$ | | Chiou et al. (1980) | ? | 79 |
| 2,3-dichlorophenol $C_6H_4Cl_2O$ [576-24-9] UMPSXRYVXUPCOS-UHFFFAOYSA-N | 2.9 | | HSDB (2015) | V | |
| 2,4-dichlorophenol $C_6H_4Cl_2O$ [120-83-2] HFZWRUODUSTPEG-UHFFFAOYSA-N | 3.4 | | Sheikheldin et al. (2001) | M | 12 |
| | 6.6 | 6800 | Tabai et al. (1997) | M | 11 |
| | 2.3 | | Duchowicz et al. (2020) | V | 186 |
| | 2.8 | | HSDB (2015) | V | |
| | 2.3 | | Mackay et al. (2006c) | V | |
| | 2.3 | | Mackay et al. (1995) | V | |
| | 2.3 | | Shiu et al. (1994) | V | |
| | 9.0 | | Leuenberger et al. (1985) | V | 416 |
| | 1.5 | | Goldstein (1982) | X | 446 |
| | 1.5 | 4900 | Goldstein (1982) | X | 298 |
| | 1.8 | | Ryan et al. (1988) | C | |
| | $2.2\times10^1$ | | Duchowicz et al. (2020) | Q | |
| | $3.2\times10^1$ | | Zhang et al. (2010) | Q | 287, 288 |
| | 8.0 | | Zhang et al. (2010) | Q | 287, 289 |
| | 1.1 | | Zhang et al. (2010) | Q | 287, 290 |
| | 4.6 | | Zhang et al. (2010) | Q | 287, 291 |
| | 8.2 | | Hilal et al. (2008) | Q | |
| | 4.4 | | Modarresi et al. (2007) | Q | 67 |
| | | 6300 | Kühne et al. (2005) | Q | |
| | | 7400 | Kühne et al. (2005) | ? | |
| 2,5-dichlorophenol $C_6H_4Cl_2O$ [583-78-8] RANCECPPZPIPNO-UHFFFAOYSA-N | 1.6 | | HSDB (2015) | V | |
| 2,6-dichlorophenol $C_6H_4Cl_2O$ [87-65-0] HOLHYSJJBXSLMV-UHFFFAOYSA-N | 3.7 | | Duchowicz et al. (2020) | V | 186 |
| | 3.7 | | HSDB (2015) | V | |
| | 1.3 | | Mackay et al. (2006c) | V | |
| | 3.3 | | Mackay et al. (1995) | V | |
| | 1.2 | | Duchowicz et al. (2020) | Q | |
| | 3.8 | | Yaffe et al. (2003) | Q | 248, 249 |
| | $2.5\times10^1$ | | Katritzky et al. (1998) | Q | |
| 3,4-dichlorophenol $C_6H_4Cl_2O$ [95-77-2] WDNBURPWRNALGP-UHFFFAOYSA-N | $2.1\times10^1$ | | HSDB (2015) | Q | 99 |





Table A6.4: Oxygenated chlorocarbons (C, H, O, Cl) (. . . continued)

| Substance<br>Formula<br>(Trivial Name)<br>[CAS Registry Number]<br>InChIKey | $H_s^{cp}$<br>(at $T^\ominus$)<br>$\left[\dfrac{\text{mol}}{\text{m}^3\,\text{Pa}}\right]$ | $\dfrac{\text{d}\ln H_s^{cp}}{\text{d}(1/T)}$<br><br>[K] | Reference | Type | Note |
|---|---|---|---|---|---|
| 3,5-dichlorophenol<br>$C_6H_4Cl_2O$<br>[591-35-5]<br>VPOMSPZBQMDLTM-UHFFFAOYSA-N | $4.0\times10^1$<br>$4.1\times10^1$<br>$1.8\times10^2$<br>$4.6\times10^1$<br>$1.4\times10^1$ | | Duchowicz et al. (2020)<br>HSDB (2015)<br>Duchowicz et al. (2020)<br>Hilal et al. (2008)<br>Modarresi et al. (2007) | V<br>V<br>Q<br>Q<br>Q | 186<br><br><br><br>67 |
| 2,3,4-trichlorophenol<br>$C_6H_3Cl_3O$<br>[15950-66-0]<br>HSQFVBWFPBKHEB-UHFFFAOYSA-N | 2.5<br>2.5 | | Mackay et al. (2006c)<br>Mackay et al. (1995) | V<br>V | |
| 2,3,5-trichlorophenol<br>$C_6H_3Cl_3O$<br>[933-78-8]<br>WWGQHTJIFOQAOC-UHFFFAOYSA-N | 2.5<br>2.5 | | Mackay et al. (2006c)<br>Mackay et al. (1995) | V<br>V | |
| 2,4,5-trichlorophenol<br>$C_6H_3Cl_3O$<br>[95-95-4]<br>LHJGJYXLEPZJPM-UHFFFAOYSA-N | 6.1<br>6.2<br>1.9<br>$4.6\times10^{-1}$<br>1.9<br>7.6<br>$4.6\times10^1$<br>$2.0\times10^1$<br>5.7 | | Duchowicz et al. (2020)<br>HSDB (2015)<br>Mackay et al. (2006c)<br>Fogg and Sangster (2003)<br>Mackay et al. (1995)<br>Leuenberger et al. (1985)<br>Duchowicz et al. (2020)<br>Hilal et al. (2008)<br>Modarresi et al. (2007) | V<br>V<br>V<br>V<br>V<br>V<br>Q<br>Q<br>Q | 186<br><br><br><br><br>416<br><br><br>67 |
| 2,3,6-trichlorophenol<br>$C_6H_3Cl_3O$<br>[933-75-5]<br>XGCHAIDDPMFRLJ-UHFFFAOYSA-N | $4.3\times10^1$ | | HSDB (2015) | Q | 99 |
| 2,4,6-trichlorophenol<br>$C_6H_3Cl_3O$<br>[88-06-2]<br>LINPIYWFGCPVIE-UHFFFAOYSA-N | $1.9\times10^1$<br>2.0<br>3.8<br>3.8<br>1.8<br>$1.6\times10^2$<br>1.8<br>7.6<br>1.4<br>$1.6\times10^1$<br>2.4<br>2.5<br>$4.3\times10^1$<br>$2.8\times10^{-2}$<br>$8.8\times10^{-1}$<br>$9.7\times10^{-1}$<br>2.2<br>3.3<br> | <br><br><br><br><br><br><br><br>5000<br><br><br><br><br><br><br><br><br><br>6400 | Chao et al. (2017)<br>Yoshida et al. (1987)<br>Duchowicz et al. (2020)<br>HSDB (2015)<br>Mackay et al. (2006c)<br>Lide and Frederikse (1995)<br>Mackay et al. (1995)<br>Leuenberger et al. (1985)<br>Goldstein (1982)<br>Howard (1989)<br>Ryan et al. (1988)<br>Duchowicz et al. (2020)<br>Zhang et al. (2010)<br>Zhang et al. (2010)<br>Zhang et al. (2010)<br>Zhang et al. (2010)<br>Hilal et al. (2008)<br>Modarresi et al. (2007)<br>Kühne et al. (2005) | M<br>M<br>V<br>V<br>V<br>V<br>V<br>V<br>X<br>X<br>C<br>Q<br>Q<br>Q<br>Q<br>Q<br>Q<br>Q<br>Q | <br>730, 12<br>186<br><br><br><br><br>416<br>298<br>418<br><br><br>287, 288<br>287, 289<br>287, 290<br>287, 291<br><br>67<br> |





Table A6.4: Oxygenated chlorocarbons (C, H, O, Cl) (. . . continued)

| Substance Formula (Trivial Name) [CAS Registry Number] InChIKey | $H_s^{cp}$ (at $T^{\ominus}$) $\left[\dfrac{\text{mol}}{\text{m}^3\,\text{Pa}}\right]$ | $\dfrac{\text{d}\ln H_s^{cp}}{\text{d}(1/T)}$ [K] | Reference | Type | Note |
|---|---|---|---|---|---|
| | 3.8 | | Yaffe et al. (2003) | Q | 248, 249 |
| | $1.8\times10^1$ | | Katritzky et al. (1998) | Q | |
| | | 6500 | Kühne et al. (2005) | ? | |
| 3,4,5-trichlorophenol C$_6$H$_3$Cl$_3$O [609-19-8] GBNHEBQXJVDXSW-UHFFFAOYSA-N | $4.3\times10^1$ | | HSDB (2015) | Q | 99 |
| 2,3,4,5-tetrachlorophenol C$_6$H$_2$Cl$_4$O [4901-51-3] RULKYXXCCZZKDZ-UHFFFAOYSA-N | 7.2 7.2 $2.8\times10^1$ | | Mackay et al. (2006c) Mackay et al. (1995) HSDB (2015) | V V Q | 99 |
| 2,3,4,6-tetrachlorophenol C$_6$H$_2$Cl$_4$O [58-90-2] VGVRPFIJEJYOFN-UHFFFAOYSA-N | 1.1 7.6 2.8 2.8 6.6 $5.8\times10^1$ $4.1\times10^{-2}$ 3.9 3.1 | | Duchowicz et al. (2020) HSDB (2015) Mackay et al. (2006c) Mackay et al. (1995) Duchowicz et al. (2020) Zhang et al. (2010) Zhang et al. (2010) Zhang et al. (2010) Zhang et al. (2010) | V V V V Q Q Q Q Q | 186 287, 288 287, 289 287, 290 287, 291 |
| 2,3,5,6-tetrachlorophenol C$_6$H$_2$Cl$_4$O [935-95-5] KEWNKZNZRIAIAK-UHFFFAOYSA-N | 4.3 4.3 $2.8\times10^1$ | | Mackay et al. (2006c) Mackay et al. (1995) HSDB (2015) | V V Q | 99 |
| hydroxypentachlorobenzene C$_6$HCl$_5$O (pentachlorophenol) [87-86-5] IZUPBVBPLAPZRR-UHFFFAOYSA-N | $4.1\times10^2$ $1.3\times10^1$ $1.1\times10^{-2}$ $1.3\times10^1$ $2.3\times10^1$ $2.3\times10^1$ $2.2\times10^{-1}$ $1.1\times10^{-1}$ 4.7 3.4 $7.9\times10^1$ $6.0\times10^{-2}$ 6.5 4.0 $2.5\times10^{-1}$ $1.3\times10^1$ $7.9\times10^1$ 1.8 | 1300 7800 7400 | Hellmann (1987) Mackay et al. (2006c) Mackay et al. (2006d) Fogg and Sangster (2003) Mackay et al. (1995) Riederer (1990) Suntio et al. (1988) Barcelo and Hennion (1997) Goldstein (1982) McCarty (1980) Ryan et al. (1988) Zhang et al. (2010) Zhang et al. (2010) Zhang et al. (2010) Zhang et al. (2010) Goodarzi et al. (2010) Modarresi et al. (2007) Kühne et al. (2005) Meylan and Howard (1991) Fogg and Sangster (2003) Kühne et al. (2005) | M V V V V V V X X X C Q Q Q Q Q Q Q Q E ? | 87 558 12 567 298 368 287, 288 287, 289 287, 290 287, 291 568, 569 67 |



Table A6.4: Oxygenated chlorocarbons (C, H, O, Cl) (...continued)

| Substance<br>Formula<br>(Trivial Name)<br>[CAS Registry Number]<br>InChIKey | $H_s^{cp}$<br>(at $T^{\ominus}$)<br>$\left[\dfrac{\text{mol}}{\text{m}^3\,\text{Pa}}\right]$ | $\dfrac{\text{d}\ln H_s^{cp}}{\text{d}(1/T)}$<br><br>[K] | Reference | Type | Note |
|---|---|---|---|---|---|
| 3,4,5-trichloro-1,2-benzenediol<br>$C_6H_3Cl_3O_2$<br>(3,4,5-trichlorocatechol)<br>[56961-20-7]<br>FUTDYIMYZIMPBJ-UHFFFAOYSA-N | $2.4\times10^2$ | | Lei et al. (1999) | V | |
| 4,5-dichloro-1,2-benzenediol<br>$C_6H_4Cl_2O_2$<br>(4,5-dichlorocatechol)<br>[3428-24-8]<br>ACCHWUWBKYGKNM-UHFFFAOYSA-N | $1.3\times10^3$ | | Lei et al. (1999) | V | |
| 3,4,5,6-tetrachloro-1,2-benzenediol<br>$C_6H_2Cl_4O_2$<br>(tetrachlorocatechol)<br>[1198-55-6]<br>RRBMVWQICIXSEO-UHFFFAOYSA-N | $2.9\times10^1$ | | Lei et al. (1999) | V | |
| 2,3,5,6-tetrachloro-$p$-benzoquinone<br>$C_6Cl_4O_2$<br>(chloranil)<br>[118-75-2]<br>UGNWTBMOAKPKBL-UHFFFAOYSA-N | $1.5\times10^3$ | | HSDB (2015) | V | |
| 2-chloro-5-methylphenol<br>$C_7H_7ClO$<br>[615-74-7]<br>SMFHPCZZAAMJJO-UHFFFAOYSA-N | $2.1\times10^1$ | | HSDB (2015) | Q | 99 |
| 4-chloro-2-methylphenol<br>$C_7H_7ClO$<br>[1570-64-5]<br>RHPUJHQBPORFGV-UHFFFAOYSA-N | 8.7<br>9.0<br>$8.5\times10^1$<br>$1.6\times10^1$<br>1.5 | | Duchowicz et al. (2020)<br>Woodrow et al. (1990)<br>Duchowicz et al. (2020)<br>Hilal et al. (2008)<br>Modarresi et al. (2007) | V<br>V<br>Q<br>Q<br>Q | 186<br><br><br><br>67 |
| 4-chloro-3-methylphenol<br>$C_7H_7ClO$<br>[59-50-7]<br>CFKMVGJGLGKFKI-UHFFFAOYSA-N | 4.0<br>4.1<br>$3.9\times10^1$<br>4.0<br>$8.5\times10^1$<br>$2.2\times10^1$<br>$1.3\times10^1$<br>$2.8\times10^1$<br>$9.2\times10^1$<br>$1.2\times10^1$<br>8.8<br>$3.9\times10^1$<br>$1.7\times10^1$<br>$1.3\times10^2$ | | Duchowicz et al. (2020)<br>HSDB (2015)<br>Abraham et al. (1994a)<br>Ryan et al. (1988)<br>Duchowicz et al. (2020)<br>Zhang et al. (2010)<br>Zhang et al. (2010)<br>Zhang et al. (2010)<br>Zhang et al. (2010)<br>Hilal et al. (2008)<br>Modarresi et al. (2007)<br>Yaffe et al. (2003)<br>English and Carroll (2001)<br>Nirmalakhandan et al. (1997) | V<br>V<br>R<br>C<br>Q<br>Q<br>Q<br>Q<br>Q<br>Q<br>Q<br>Q<br>Q<br>Q | 186<br><br><br><br><br>287, 288<br>287, 289<br>287, 290<br>287, 291<br><br>67<br>248, 249<br>230, 231<br> |



Table A6.4: Oxygenated chlorocarbons (C, H, O, Cl) (...continued)

| Substance<br>Formula<br>(Trivial Name)<br>[CAS Registry Number]<br>InChIKey | $H_s^{cp}$<br>(at $T^{\ominus}$)<br>$\left[\dfrac{\text{mol}}{\text{m}^3\,\text{Pa}}\right]$ | $\dfrac{\text{d}\ln H_s^{cp}}{\text{d}(1/T)}$<br><br>[K] | Reference | Type | Note |
|---|---|---|---|---|---|
| 1-chloro-2-methoxybenzene<br>$C_7H_7ClO$<br>(2-chloroanisole)<br>[766-51-8]<br>QGRPVMLBTFGQDQ-UHFFFAOYSA-N | $1.0\times10^{-1}$<br>$1.1\times10^{-1}$<br>$1.8\times10^{-1}$ | | Pfeifer et al. (2001)<br>Duchowicz et al. (2020)<br>Duchowicz et al. (2020) | M<br>V<br>Q | 731<br>186 |
| 1-chloro-3-methoxybenzene<br>$C_7H_7ClO$<br>(3-chloroanisole)<br>[2845-89-8]<br>YUKILTJWFRTXGB-UHFFFAOYSA-N | $4.5\times10^{-2}$ | | Pfeifer et al. (2001) | M | 731 |
| 1-chloro-4-methoxybenzene<br>$C_7H_7ClO$<br>(4-chloroanisole)<br>[623-12-1]<br>YRGAYAGBVIXNAQ-UHFFFAOYSA-N | $5.8\times10^{-2}$ | | Pfeifer et al. (2001) | M | 731 |
| 1,2-dichloro-3-methoxybenzene<br>$C_7H_6Cl_2O$<br>(2,3-dichloroanisole)<br>[1984-59-4]<br>HFEASCCDHUVYKU-UHFFFAOYSA-N | $2.2\times10^{-2}$ | | Pfeifer et al. (2001) | M | 731 |
| 1,5-dichloro-2-methoxybenzene<br>$C_7H_6Cl_2O$<br>(2,4-dichloroanisole)<br>[553-82-2]<br>CICQUFBZCADHHX-UHFFFAOYSA-N | $1.2\times10^{-2}$ | | Pfeifer et al. (2001) | M | 731 |
| 1,4-dichloro-2-methoxybenzene<br>$C_7H_6Cl_2O$<br>(2,5-dichloroanisole)<br>[1984-58-3]<br>QKMNFFSBZRGHDJ-UHFFFAOYSA-N | $2.1\times10^{-2}$<br>$5.7\times10^{-2}$<br>$1.4\times10^{-2}$<br>$1.4\times10^{-1}$<br>$4.8\times10^{-2}$ | | Pfeifer et al. (2001)<br>Zhang et al. (2010)<br>Zhang et al. (2010)<br>Zhang et al. (2010)<br>Zhang et al. (2010) | M<br>Q<br>Q<br>Q<br>Q | 731<br>287, 288<br>287, 289<br>287, 290<br>287, 291 |
| 1,3-dichloro-2-methoxybenzene<br>$C_7H_6Cl_2O$<br>(2,6-dichloroanisole)<br>[1984-65-2]<br>KZLMCDNAVVJKPX-UHFFFAOYSA-N | $8.8\times10^{-3}$ | | Pfeifer et al. (2001) | M | 731 |
| 1,2-dichloro-4-methoxybenzene<br>$C_7H_6Cl_2O$<br>(3,4-dichloroanisole)<br>[36404-30-5]<br>VISJRVXHPNMYRH-UHFFFAOYSA-N | $9.2\times10^{-3}$ | | Pfeifer et al. (2001) | M | 731 |



Table A6.4: Oxygenated chlorocarbons (C, H, O, Cl) (. . . continued)

| Substance<br>Formula<br>(Trivial Name)<br>[CAS Registry Number]<br>InChIKey | $H_s^{cp}$<br>(at $T^{\ominus}$)<br>$\left[\dfrac{\mathrm{mol}}{\mathrm{m^3\,Pa}}\right]$ | $\dfrac{\mathrm{d}\ln H_s^{cp}}{\mathrm{d}(1/T)}$<br><br>[K] | Reference | Type | Note |
|---|---|---|---|---|---|
| 1,3-dichloro-5-methoxybenzene<br>$C_7H_6Cl_2O$<br>(3,5-dichloroanisole)<br>[33719-74-3]<br>SSNXYMVLSOMJLU-UHFFFAOYSA-N | $2.3\times10^{-3}$ | | Pfeifer et al. (2001) | M | 731 |
| 1,2,3-trichloro-4-methoxybenzene<br>$C_7H_5Cl_3O$<br>(2,3,4-trichloroanisole)<br>[54135-80-7]<br>FRQUNVLMWIYOLV-UHFFFAOYSA-N | $1.3\times10^{-2}$ | | Pfeifer et al. (2001) | M | 731 |
| 1,2,5-trichloro-3-methoxybenzene<br>$C_7H_5Cl_3O$<br>(2,3,5-trichloroanisole)<br>[54135-81-8]<br>MKERQGLKSFEKAE-UHFFFAOYSA-N | $7.6\times10^{-3}$ | | Pfeifer et al. (2001) | M | 731 |
| 1,2,4-trichloro-3-methoxybenzene<br>$C_7H_5Cl_3O$<br>(2,3,6-trichloroanisole)<br>[50375-10-5]<br>OTFNCXLUCRUNCH-UHFFFAOYSA-N | $1.1\times10^{-2}$<br>$9.8\times10^{-3}$<br>$1.8\times10^{-2}$<br>$2.0\times10^{-2}$<br>$7.6\times10^{-2}$ | 4500 | Diaz et al. (2005)<br>Pfeifer et al. (2001)<br>Hilal et al. (2008)<br>Modarresi et al. (2007)<br>Meylan and Howard (1991) | M<br>M<br>Q<br>Q<br>Q | <br>731<br><br>67<br> |
| 1,2,4-trichloro-5-methoxybenzene<br>$C_7H_5Cl_3O$<br>(2,4,5-trichloroanisole)<br>[6130-75-2]<br>SXKBHOQOOGRFJF-UHFFFAOYSA-N | $1.1\times10^{-2}$ | | Pfeifer et al. (2001) | M | 731 |
| 1,3,5-trichloro-2-methoxybenzene<br>$C_7H_5Cl_3O$<br>(2,4,6-trichloroanisole)<br>[87-40-1]<br>WCVOGSZTONGSQY-UHFFFAOYSA-N | $2.3\times10^{-2}$<br>$4.4\times10^{-3}$<br>$4.6\times10^{-3}$ | 5500<br>640 | Wu et al. (2022a)<br>Diaz et al. (2005)<br>Pfeifer et al. (2001) | M<br>M<br>M | <br><br>731 |
| 1,2,3-trichloro-5-methoxybenzene<br>$C_7H_5Cl_3O$<br>(3,4,5-trichloroanisole)<br>[54135-82-9]<br>GUCFBWGWRCILHN-UHFFFAOYSA-N | $4.4\times10^{-3}$ | | Pfeifer et al. (2001) | M | 731 |
| 1,2,3,4-tetrachloro-5-methoxybenzene<br>$C_7H_4Cl_4O$<br>(2,3,4,5-tetrachloroanisole)<br>[938-86-3]<br>FUUHMSUPRUNWRQ-UHFFFAOYSA-N | $6.5\times10^{-3}$ | | Pfeifer et al. (2001) | M | 731 |





Table A6.4: Oxygenated chlorocarbons (C, H, O, Cl) (...continued)

| Substance Formula (Trivial Name) [CAS Registry Number] InChIKey | $H_s^{cp}$ (at $T^{\ominus}$) $\left[\dfrac{\text{mol}}{\text{m}^3\,\text{Pa}}\right]$ | $\dfrac{\text{d}\ln H_s^{cp}}{\text{d}(1/T)}$ [K] | Reference | Type | Note |
|---|---|---|---|---|---|
| 1,2,3,5-tetrachloro-4-methoxybenzene $C_7H_4Cl_4O$ (2,3,4,6-tetrachloroanisole) [938-22-7] ITXDBGLYYSJNPK-UHFFFAOYSA-N | $3.1\times10^{-3}$ | | Pfeifer et al. (2001) | M | 731 |
| 1,2,4,5-tetrachloro-3-methoxybenzene $C_7H_4Cl_4O$ (2,3,5,6-tetrachloroanisole) [6936-40-9] WMMFIDNWZNCBCT-UHFFFAOYSA-N | $3.2\times10^{-3}$ | | Pfeifer et al. (2001) | M | 731 |
| pentachloromethoxybenzene $C_7H_3Cl_5O$ (pentachloroanisole) [1825-21-4] BBABSCYTNHOKOG-UHFFFAOYSA-N | $2.1\times10^{-3}$ $5.1\times10^{-3}$ | | Pfeifer et al. (2001) HSDB (2015) | M Q | 731 99 |
| 4,5-dichloro-2-methoxyphenol $C_7H_6Cl_2O_2$ (4,5-dichloroguaiacol) [2460-49-3] HAAFFTHBNFBVKY-UHFFFAOYSA-N | 5.2 2.3 | | Mackay et al. (2006c) Lei et al. (1999) | V V | |
| 3,4,5-trichloro-2-methoxyphenol $C_7H_5Cl_3O_2$ (3,4,5-trichloroguaiacol) [57057-83-7] RKEHLKXRUVUBJN-UHFFFAOYSA-N | $1.1\times10^1$ 8.3 $9.7\times10^1$ | | Duchowicz et al. (2020) Mackay et al. (2006c) Lei et al. (1999) Duchowicz et al. (2020) | V V V Q | 186 420 |
| 4,5,6-trichloro-2-methoxyphenol $C_7H_5Cl_3O_2$ (4,5,6-trichloroguaiacol) [2668-24-8] NIAJPNQTKGWEOI-UHFFFAOYSA-N | 7.4 7.1 | | Mackay et al. (2006c) Lei et al. (1999) | V V | |
| 2,3,4,5-tetrachloro-6-methoxyphenol $C_7H_4Cl_4O_2$ (tetrachloroguaiacol) [2539-17-5] YZZVKLJKDFFSFL-UHFFFAOYSA-N | 1.4 6.2 6.7 8.4 | | Duchowicz et al. (2020) Mackay et al. (2006c) Lei et al. (1999) Duchowicz et al. (2020) | V V V Q | 186 |
| 2-chlorobenzoic acid $C_7H_5ClO_2$ [118-91-2] IKCLCGXPQILATA-UHFFFAOYSA-N | $1.5\times10^2$ $1.3\times10^2$ $3.5\times10^2$ $1.4\times10^2$ | | Duchowicz et al. (2020) Yaws (2003) Duchowicz et al. (2020) Gharagheizi et al. (2010) | V X Q Q | 186 237 246 |





Table A6.4: Oxygenated chlorocarbons (C, H, O, Cl) (. . . continued)

| Substance Formula (Trivial Name) [CAS Registry Number] InChIKey | $H_s^{cp}$ (at $T^\ominus$) $\left[\dfrac{\mathrm{mol}}{\mathrm{m^3\,Pa}}\right]$ | $\dfrac{\mathrm{d\ln} H_s^{cp}}{\mathrm{d}(1/T)}$ [K] | Reference | Type | Note |
|---|---|---|---|---|---|
| 3-chlorobenzoic acid $C_7H_5ClO_2$ [535-80-8] LULAYUGMBFYYEX-UHFFFAOYSA-N | $5.7\times10^1$ $2.5\times10^2$ | | Abraham et al. (2019) HSDB (2015) | Q Q | 545 |
| 4-chlorobenzoic acid $C_7H_5ClO_2$ [74-11-3] XRHGYUZYPHTUJZ-UHFFFAOYSA-N | $2.5\times10^1$ | | Abraham et al. (2019) | Q | |
| 3,4-dichlorobenzoic acid $C_7H_4Cl_2O_2$ [51-44-5] VPHHJAOJUJHJKD-UHFFFAOYSA-N | $2.1\times10^1$ | | Abraham et al. (2019) | Q | |
| 2,3,6-trichlorobenzoic acid $C_7H_3Cl_3O_2$ [50-31-7] XZIDTOHMJBOSOX-UHFFFAOYSA-N | $4.7\times10^2$ $6.5\times10^2$ | | Duchowicz et al. (2020) Duchowicz et al. (2020) | V Q | 186 |
| 1,2,3-trichloro-4,5-dimethoxybenzene $C_8H_7Cl_3O_2$ (3,4,5-trichloroveratrole) [16766-29-3] VKNITLPENCJQOP-UHFFFAOYSA-N | $2.7\times10^{-1}$ | | Lei et al. (1999) | V | |
| 1,2,3,4-tetrachloro-5,6-dimethoxybenzene $C_8H_6Cl_4O_2$ (tetrachloroveratrole) [944-61-6] NCYHCGGUQGDEQW-UHFFFAOYSA-N | $9.1\times10^{-2}$ 1.7 2.0 2.6 $7.5\times10^{-1}$ | | Lei et al. (1999) Zhang et al. (2010) Zhang et al. (2010) Zhang et al. (2010) Zhang et al. (2010) | V Q Q Q Q | 287, 288 287, 289 287, 290 287, 291 |
| 3-chloro-2,6-dimethoxyphenol $C_8H_9ClO_3$ (3-chlorosyringol) [18113-22-9] WYEMCZZCZXKDNC-UHFFFAOYSA-N | $4.2\times10^1$ | | Lei et al. (1999) | V | |
| 3,5-dichloro-2,6-dimethoxyphenol $C_8H_8Cl_2O_3$ (3,5-dichlorosyringol) [78782-46-4] IDKMFABKPPHDBI-UHFFFAOYSA-N | $1.4\times10^1$ | | Lei et al. (1999) | V | |
| 3,5-dichloro-2-hydroxybenzoic acid $C_7H_4Cl_2O_3$ [320-72-9] CNJGWCQEGROXEE-UHFFFAOYSA-N | $1.3\times10^3$ $4.3\times10^2$ $7.5\times10^4$ $2.2\times10^2$ | | Zhang et al. (2010) Zhang et al. (2010) Zhang et al. (2010) Zhang et al. (2010) | Q Q Q Q | 287, 288 287, 289 287, 290 287, 291 |





Table A6.4: Oxygenated chlorocarbons (C, H, O, Cl) (...continued)

| Substance Formula (Trivial Name) [CAS Registry Number] InChIKey | $H_s^{cp}$ (at $T^{\ominus}$) $\left[\dfrac{\text{mol}}{\text{m}^3\,\text{Pa}}\right]$ | $\dfrac{\text{d}\ln H_s^{cp}}{\text{d}(1/T)}$ [K] | Reference | Type | Note |
|---|---|---|---|---|---|
| 3,6-dichloro-2-hydroxybenzoic acid | $1.3\times10^3$ | | Zhang et al. (2010) | Q | 287, 288 |
| $C_7H_4Cl_2O_3$ | $3.4\times10^3$ | | Zhang et al. (2010) | Q | 287, 289 |
| [3401-80-7] | $1.1\times10^1$ | | Zhang et al. (2010) | Q | 287, 290 |
| FKIKPQHMWFZFEB-UHFFFAOYSA-N | $2.2\times10^2$ | | Zhang et al. (2010) | Q | 287, 291 |
| 5-chloro-2-methoxybenzoic acid | $2.1\times10^3$ | | Zhang et al. (2010) | Q | 287, 288 |
| $C_8H_7ClO_3$ | $3.8\times10^1$ | | Zhang et al. (2010) | Q | 287, 289 |
| [3438-16-2] | $9.5\times10^3$ | | Zhang et al. (2010) | Q | 287, 290 |
| HULDRQRKKXRXBI-UHFFFAOYSA-N | $6.0\times10^3$ | | Zhang et al. (2010) | Q | 287, 291 |
| 2-chloroacetophenone | 2.8 | | HSDB (2015) | Q | 99 |
| $C_8H_7ClO$ | | | | | |
| [532-27-4] | | | | | |
| IMACFCSSMIZSPP-UHFFFAOYSA-N | | | | | |
| 2,2,2',4',5'-pentachloroacetophenone | $2.0\times10^1$ | | Zhang et al. (2010) | Q | 287, 288 |
| $C_8H_3Cl_5O$ | 5.7 | | Zhang et al. (2010) | Q | 287, 289 |
| [1203-86-7] | 1.0 | | Zhang et al. (2010) | Q | 287, 290 |
| WKJXVVFAALGBOH-UHFFFAOYSA-N | 6.0 | | Zhang et al. (2010) | Q | 287, 291 |
| tetrachloroterephthaloyl chloride | $1.0\times10^1$ | | Zhang et al. (2010) | Q | 287, 288 |
| $C_8Cl_6O_2$ | $1.9\times10^1$ | | Zhang et al. (2010) | Q | 287, 289 |
| [719-32-4] | $3.4\times10^{-2}$ | | Zhang et al. (2010) | Q | 287, 290 |
| YJIRZJAZKDWEIJ-UHFFFAOYSA-N | $1.3\times10^3$ | | Zhang et al. (2010) | Q | 287, 291 |
| chloroxylenol | $1.9\times10^1$ | | HSDB (2015) | Q | 99 |
| $C_8H_9ClO$ | $1.9\times10^1$ | | Zhang et al. (2010) | Q | 287, 288 |
| [88-04-0] | $1.5\times10^1$ | | Zhang et al. (2010) | Q | 287, 289 |
| OSDLLIBGSJNGJE-UHFFFAOYSA-N | $2.0\times10^1$ | | Zhang et al. (2010) | Q | 287, 290 |
| | $5.1\times10^1$ | | Zhang et al. (2010) | Q | 287, 291 |
| 4,5,6,7-tetrachloro-1,3-isobenzofurandione | 5.2 | | Zhang et al. (2010) | Q | 287, 288 |
| $C_8Cl_4O_3$ | $1.8\times10^4$ | | Zhang et al. (2010) | Q | 287, 289 |
| [117-08-8] | $1.9\times10^2$ | | Zhang et al. (2010) | Q | 287, 290 |
| AUHHYELHRWCWEZ-UHFFFAOYSA-N | $4.3\times10^1$ | | Zhang et al. (2010) | Q | 287, 291 |
| 3,4,5-trichloro-2,6-dimethoxyphenol $C_8H_7Cl_3O_3$ (trichlorosyringol) [2539-26-6] ZZCSBXFJFLSDRR-UHFFFAOYSA-N | $4.5\times10^1$ | | Lei et al. (1999) | V | |
| 4,5,6,7-tetrachlorophthalide | $1.8\times10^1$ | | Kawamoto and Urano (1989) | M | |
| $C_8H_2Cl_4O_2$ | $3.1\times10^3$ | | Duchowicz et al. (2020) | V | 186 |
| [27355-22-2] | 8.6 | | Duchowicz et al. (2020) | Q | |
| NMWKWBPNKPGATC-UHFFFAOYSA-N | | | | | |



Table A6.4: Oxygenated chlorocarbons (C, H, O, Cl) (...continued)

| Substance / Formula / (Trivial Name) / [CAS Registry Number] / InChIKey | $H_s^{cp}$ (at $T^\ominus$) $\left[\dfrac{\mathrm{mol}}{\mathrm{m^3\,Pa}}\right]$ | $\dfrac{\mathrm{d}\ln H_s^{cp}}{\mathrm{d}(1/T)}$ [K] | Reference | Type | Note |
|---|---|---|---|---|---|
| dicamba | $4.5\times10^3$ | | Duchowicz et al. (2020) | V | 186 |
| $C_8H_6Cl_2O_3$ | $2.3\times10^4$ | | HSDB (2015) | V | |
| (banvel) | $4.5\times10^3$ | | Mackay et al. (2006d) | V | |
| [1918-00-9] | $8.3\times10^3$ | | Suntio et al. (1988) | V | 12 |
| IWEDIXLBFLAXBO-UHFFFAOYSA-N | $8.2\times10^1$ | | Barcelo and Hennion (1997) | X | 567 |
| | $2.2\times10^4$ | | Armbrust (2000) | C | |
| | $5.9\times10^3$ | | Duchowicz et al. (2020) | Q | |
| | $1.2$ | | Goodarzi et al. (2010) | Q | 568, 569 |
| | $1.0\times10^4$ | | Maniere et al. (2011) | ? | 165 |
| (2,4-dichlorophenoxy)-ethanoic acid | $1.4\times10^{-1}$ | | Rice et al. (1997b) | M | 732, 12 |
| $C_8H_6Cl_2O_3$ | $1.2$ | | Rice et al. (1997b) | M | 732, 12 |
| ((2,4-dichlorophenoxy)-acetic acid; 2,4-D) | $2.8\times10^2$ | | Duchowicz et al. (2020) | V | 186 |
| [94-75-7] | $5.0\times10^4$ | | Mackay et al. (2006c) | V | |
| OVSKIKFHRZPJSS-UHFFFAOYSA-N | $2.3\times10^4$ | | Mackay et al. (2006d) | V | |
| | $4.0\times10^3$ | | Mackay et al. (2006d) | V | |
| | $2.9\times10^2$ | | Mackay et al. (1995) | V | |
| | $1.8$ | | Riederer (1990) | V | |
| | $1.8$ | | Suntio et al. (1988) | V | 12 |
| | $7.2\times10^4$ | | Howard (1991) | X | 412 |
| | $9.7\times10^2$ | | Howard (1991) | X | 412 |
| | $5.5\times10^6$ | | Armbrust (2000) | C | |
| | $8.7\times10^2$ | | Duchowicz et al. (2020) | Q | |
| | $2.5\times10^5$ | | Maniere et al. (2011) | ? | 12, 165 |
| 2,4,5-trichlorophenoxyethanoic acid | $1.7\times10^2$ | | Mackay et al. (2006d) | V | |
| $C_8H_5Cl_3O_3$ | $1.7\times10^2$ | | Riederer (1990) | V | |
| (2,4,5-T) | $1.7\times10^2$ | | Suntio et al. (1988) | V | 12 |
| [93-76-5] | $8.4\times10^5$ | | MacBean (2012a) | ? | |
| SMYMJHWAQXWPDB-UHFFFAOYSA-N | | | | | |
| 1,4-dichloro-2,5-dimethoxybenzene | $9.7\times10^{-2}$ | | Duchowicz et al. (2020) | V | 186 |
| $C_8H_8Cl_2O_2$ | $9.9\times10^{-2}$ | | HSDB (2015) | V | |
| [2675-77-6] | | | Mackay et al. (2006d) | V | 558 |
| PFIADAMVCJPXSF-UHFFFAOYSA-N | $9.4\times10^{-1}$ | | Duchowicz et al. (2020) | Q | |
| 2,3,6-trichlorophenylacetic acid | $8.3\times10^{-1}$ | | Mackay et al. (2006d) | V | |
| $C_8H_5Cl_3O_2$ | $5.5\times10^2$ | | HSDB (2015) | Q | 99 |
| [85-34-7] | | | | | |
| QZXCCPZJCKEPSA-UHFFFAOYSA-N | | | | | |



Table A6.4: Oxygenated chlorocarbons (C, H, O, Cl) (... continued)

| Substance<br>Formula<br>(Trivial Name)<br>[CAS Registry Number]<br>InChIKey | $H_s^{cp}$<br>(at $T^\ominus$)<br>$\left[\dfrac{\mathrm{mol}}{\mathrm{m^3\,Pa}}\right]$ | $\dfrac{\mathrm{d}\ln H_s^{cp}}{\mathrm{d}(1/T)}$<br><br>[K] | Reference | Type | Note |
|---|---|---|---|---|---|
| 4-methoxy-benzoyl chloride<br>$C_8H_7ClO_2$<br>($p$-anisoyl chloride)<br>[100-07-2]<br>MXMOTZIXVICDSD-UHFFFAOYSA-N | 1.3 | | HSDB (2015) | Q | 99 |
| (4-chlorophenoxy)acetic acid<br>$C_8H_7ClO_3$<br>[122-88-3]<br>SODPIMGUZLOIPE-UHFFFAOYSA-N | $1.0\times10^5$ | | Ebert et al. (2023) | ? | 316 |
| isobenzan<br>$C_9H_4Cl_8O$<br>[297-78-9]<br>LRWHHSXTGZSMSN-UHFFFAOYSA-N | $1.7\times10^2$ | | HSDB (2015) | Q | 99 |
| 2-chloro-4-hydroxy-3,5-<br>dimethoxybenzaldehyde<br>$C_9H_9ClO_4$<br>(2-chlorosyringaldehyde)<br>[76341-69-0]<br>GRIHRCLOUQZXPD-UHFFFAOYSA-N | $9.1\times10^1$ | | Lei et al. (1999) | V | |
| 2,6-dichloro-4-hydroxy-3,5-<br>dimethoxybenzaldehyde<br>$C_9H_8Cl_2O_4$<br>(2,6-dichlorosyringaldehyde)<br>[76330-06-8]<br>CTFRWEPMHUGVMM-UHFFFAOYSA-N | $2.7\times10^2$ | | Lei et al. (1999) | V | |
| methyl<br>2,4-dichlorophenoxyethanoate<br>$C_9H_8Cl_2O_3$<br>[1928-38-7]<br>HWIGZMADSFQMOI-UHFFFAOYSA-N | 3.6<br>$1.0\times10^1$<br>1.8<br>1.3 | | Duchowicz et al. (2020)<br>Duchowicz et al. (2020)<br>Hilal et al. (2008)<br>Modarresi et al. (2007) | V<br>Q<br>Q<br>Q | 186<br><br><br>67 |
| (2-methyl-4-chlorophenoxy)acetic<br>acid<br>$C_9H_9ClO_3$<br>(MCPA)<br>[94-74-6]<br>WHKUVVPPKQRRBV-UHFFFAOYSA-N | $>9.9\times10^1$<br>$7.4\times10^3$<br>$4.0\times10^4$<br>$9.9\times10^3$<br>$5.7\times10^2$<br>$1.8\times10^4$ | | Mabury and Crosby (1996)<br>Duchowicz et al. (2020)<br>Mackay et al. (2006d)<br>Woodrow et al. (1990)<br>Duchowicz et al. (2020)<br>Maniere et al. (2011) | M<br>V<br>V<br>V<br>Q<br>? | <br>186<br><br><br><br>165 |
| $\alpha$-(2,4-dichlorophenoxy)propionic<br>acid<br>$C_9H_8Cl_2O_3$<br>(dichloroprop)<br>[120-36-5]<br>MZHCENGPTKEIGP-UHFFFAOYSA-N | $3.7\times10^3$ | | Mackay et al. (2006d) | V | |



Table A6.4: Oxygenated chlorocarbons (C, H, O, Cl) (. . . continued)

| Substance Formula (Trivial Name) [CAS Registry Number] InChIKey | $H_s^{cp}$ (at $T^\ominus$) $\left[\dfrac{\text{mol}}{\text{m}^3\,\text{Pa}}\right]$ | $\dfrac{\text{d}\ln H_s^{cp}}{\text{d}(1/T)}$ [K] | Reference | Type | Note |
|---|---|---|---|---|---|
| (R)-2-(2,4-dichlorophenoxy)propanoic acid $C_9H_8Cl_2O_3$ (dichlorprop-p) [15165-67-0] MZHCENGPTKEIGP-RXMQYKEDSA-N | $4.0\times10^4$ <br> $1.8\times10^4$ | | Mackay et al. (2006d) <br> Maniere et al. (2011) | V <br> ? | <br> 241, 165 |
| 2-(2,4,5-trichlorophenoxy)propanoic acid $C_9H_7Cl_3O_3$ [93-72-1] ZLSWBLPERHFHIS-UHFFFAOYSA-N | $3.9\times10^4$ | | Mackay et al. (2006d) | V | |
| cloxyfonac $C_9H_9O_4Cl$ [6386-63-6] ZJRUTGDCLVIVRD-UHFFFAOYSA-N | $1.0\times10^5$ <br> $1.9\times10^6$ | | Duchowicz et al. (2020) <br> Duchowicz et al. (2020) | V <br> Q | 186 |
| tridiphane $C_{10}H_7Cl_5O$ [58138-08-2] IBZHOAONZVJLOB-UHFFFAOYSA-N | $1.9\times10^{-1}$ | | MacBean (2012a) | ? | |
| plifenat $C_{10}H_7O_2Cl_5$ [21757-82-4] FSGNOVKGEXRRHD-UHFFFAOYSA-N | $1.1\times10^4$ | | MacBean (2012a) | ? | |
| chlorfenprop-methyl $C_{10}H_{10}Cl_2O_2$ [14437-17-3] YJKIALIXRCSISK-UHFFFAOYSA-N | $4.4$ | | Ebert et al. (2023) | ? | 365 |
| ethyl 2,4-dichlorophenoxyethanoate $C_{10}H_{10}Cl_2O_3$ [533-23-3] JSLBZIVMVVHMDJ-UHFFFAOYSA-N | $3.0$ <br> $4.3$ <br> $1.2$ <br> $1.1$ | | Duchowicz et al. (2020) <br> Duchowicz et al. (2020) <br> Hilal et al. (2008) <br> Modarresi et al. (2007) | V <br> Q <br> Q <br> Q | 186 <br> <br> <br> 67 |
| mecoprop $C_{10}H_{11}ClO_3$ [93-65-2] WNTGYJSOUMFZEP-UHFFFAOYSA-N | $1.1\times10^4$ <br> <br> $9.0\times10^1$ <br> $9.0\times10^3$ <br> $2.6\times10^2$ <br> $1.4\times10^2$ | | Duchowicz et al. (2020) <br> Mackay et al. (2006d) <br> Barcelo and Hennion (1997) <br> Armbrust (2000) <br> Duchowicz et al. (2020) <br> Goodarzi et al. (2010) | V <br> V <br> X <br> C <br> Q <br> Q | 186 <br> 558 <br> 567 <br> <br> <br> 568 |



Table A6.4: Oxygenated chlorocarbons (C, H, O, Cl) (. . . continued)

| Substance Formula (Trivial Name) [CAS Registry Number] InChIKey | $H_s^{cp}$ (at $T^{\ominus}$) $\left[\dfrac{\mathrm{mol}}{\mathrm{m}^3\,\mathrm{Pa}}\right]$ | $\dfrac{\mathrm{d}\ln H_s^{cp}}{\mathrm{d}(1/T)}$ [K] | Reference | Type | Note |
|---|---|---|---|---|---|
| (R)-2-(4-chloro-2-methylphenoxy)propanoic acid | $1.0\times10^4$ | | Mackay et al. (2006d) | V | |
| $C_{10}H_{11}ClO_3$ (mecoprop-p) [16484-77-8] WNTGYJSOUMFZEP-SSDOTTSWSA-N | $1.8\times10^4$ | | Maniere et al. (2011) | ? | 12, 165 |
| dacthal | 4.4 | | Muir et al. (2004) | L | 367 |
| $C_{10}H_6Cl_4O_4$ | 4.5 | | Duchowicz et al. (2020) | V | 186 |
| (DCPA) | 4.5 | | HSDB (2015) | V | |
| [1861-32-1] NPOJQCVWMSKXDN-UHFFFAOYSA-N | $1.4\times10^2$ | | Duchowicz et al. (2020) | Q | |
| 4-(2,4-dichlorophenoxy)-butanoic acid | $4.3\times10^3$ | | HSDB (2015) | Q | 99 |
| $C_{10}H_{10}Cl_2O_3$ (2,4-DB) [94-82-6] YIVXMZJTEQBPQO-UHFFFAOYSA-N | $2.2\times10^5$ | | Maniere et al. (2011) | ? | 241, 165 |
| dichlone $C_{10}H_4Cl_2O_2$ [117-80-6] SVPKNMBRVBMTLB-UHFFFAOYSA-N | $9.7\times10^3$ | | HSDB (2015) | Q | 99 |
| 1-[(2,3,6-trichlorophenyl)methoxy]propan-2-ol | $2.0\times10^1$ | | Duchowicz et al. (2020) | V | 186 |
| $C_{10}H_{11}Cl_3O_2$ (tritac) [1861-44-5] LJWIIRATRWPHBA-UHFFFAOYSA-N | $1.7\times10^1$ | | Duchowicz et al. (2020) | Q | |
| (2,4-dichlorophenoxy)-acetic acid 1-methylethyl ester | 4.6 | | Duchowicz et al. (2020) | V | 186 |
| $C_{11}H_{12}Cl_2O_3$ | 4.5 | | HSDB (2015) | V | |
| [94-11-1] WHOKDONDRZNCBC-UHFFFAOYSA-N | 2.0 | | Duchowicz et al. (2020) | Q | |
| 4-(4-chloro-2-methylphenoxy)butanoic acid | $3.6\times10^3$ | | Duchowicz et al. (2020) | V | 186 |
| $C_{11}H_{13}ClO_3$ | $3.1\times10^3$ | | Mackay et al. (2006d) | V | |
| (MCPB) | $1.7\times10^3$ | | Duchowicz et al. (2020) | Q | |
| [94-81-5] LLWADFLAOKUBDR-UHFFFAOYSA-N | $3.3\times10^4$ | | Maniere et al. (2011) | ? | 241, 165 |



Table A6.4: Oxygenated chlorocarbons (C, H, O, Cl) (... continued)

| Substance<br>Formula<br>(Trivial Name)<br>[CAS Registry Number]<br>InChIKey | $H_s^{cp}$<br>(at $T^\ominus$)<br>$\left[\dfrac{\text{mol}}{\text{m}^3\,\text{Pa}}\right]$ | $\dfrac{\text{d}\ln H_s^{cp}}{\text{d}(1/T)}$<br><br>[K] | Reference | Type | Note |
|---|---|---|---|---|---|
| triclosan<br>$C_{12}H_7Cl_3O_2$<br>[3380-34-5]<br>XEFQLINVKFYRCS-UHFFFAOYSA-N | $4.7\times10^2$<br>$2.0\times10^3$<br>$5.7\times10^1$<br>$1.4\times10^3$<br>$8.2\times10^2$ | | HSDB (2015)<br>Zhang et al. (2010)<br>Zhang et al. (2010)<br>Zhang et al. (2010)<br>Zhang et al. (2010) | Q<br>Q<br>Q<br>Q<br>Q | 99<br>287, 288<br>287, 289<br>287, 290<br>287, 291 |
| monobutyl tetrachlorophthalate<br>$C_{12}H_{10}Cl_4O_4$<br>[24261-19-6]<br>WKYMTJULVAGWJM-UHFFFAOYSA-N | $2.0\times10^4$<br>$7.5\times10^2$<br>$5.1\times10^4$<br>$5.1\times10^4$ | | Zhang et al. (2010)<br>Zhang et al. (2010)<br>Zhang et al. (2010)<br>Zhang et al. (2010) | Q<br>Q<br>Q<br>Q | 287, 288<br>287, 289<br>287, 290<br>287, 291 |
| (2,4-dichlorophenoxy)-acetic acid<br>butyl ester<br>$C_{12}H_{14}Cl_2O_3$<br>[94-80-4]<br>UQMRAFJOBWOFNS-UHFFFAOYSA-N | $2.0\times10^1$<br><br>$2.0\times10^1$<br>5.7 | | Duchowicz et al. (2020)<br><br>HSDB (2015)<br>Duchowicz et al. (2020) | V<br><br>V<br>Q | 186 |
| sucralose<br>$C_{12}H_{19}Cl_3O_8$<br>[56038-13-2]<br>BAQAVOSOZGMPRM-QBMZZYIRSA-N | $2.5\times10^{13}$ | | HSDB (2015) | Q | 99 |
| endrin aldehyde<br>$C_{12}H_8Cl_6O$<br>[7421-93-4]<br>HCTWZIFNBBCVGM-UHFFFAOYSA-N | 2.4<br>2.3<br>4.8 | | Duchowicz et al. (2020)<br>HSDB (2015)<br>Duchowicz et al. (2020) | V<br>V<br>Q | 186 |
| clorophene<br>$C_{13}H_{11}ClO$<br>(4-chloro-2-benzylphenol)<br>[120-32-1]<br>NCKMMSIFQUPKCK-UHFFFAOYSA-N | $3.7\times10^3$<br>$3.7\times10^3$<br>$6.5\times10^2$ | | Duchowicz et al. (2020)<br>HSDB (2015)<br>Duchowicz et al. (2020) | V<br>V<br>Q | 186 |
| (4-chlorophenyl)phenylmethanone<br>$C_{13}H_9ClO$<br>(4-chlorobenzophenone)<br>[134-85-0]<br>UGVRJVHOJNYEHR-UHFFFAOYSA-N | 7.0 | | HSDB (2015) | Q | 99 |
| 1-(4-chlorophenyl)-4,4-dimethyl-<br>3-pentanone<br>$C_{13}H_{17}ClO$<br>[66346-01-8]<br>ILQGIJDYSLHIOX-UHFFFAOYSA-N | 1.1<br><br>$7.2\times10^{-1}$<br>4.2<br>$3.9\times10^{-1}$ | | Zhang et al. (2010)<br><br>Zhang et al. (2010)<br>Zhang et al. (2010)<br>Zhang et al. (2010) | Q<br><br>Q<br>Q<br>Q | 287, 288<br><br>287, 289<br>287, 290<br>287, 291 |
| hexachlorophene<br>$C_{13}H_6Cl_6O_2$<br>[70-30-4]<br>ACGUYXCXAPNIKK-UHFFFAOYSA-N | $1.8\times10^7$<br>$1.1\times10^7$<br>$2.5\times10^5$<br>$1.2\times10^4$<br>$6.5\times10^5$ | | HSDB (2015)<br>Zhang et al. (2010)<br>Zhang et al. (2010)<br>Zhang et al. (2010)<br>Zhang et al. (2010) | Q<br>Q<br>Q<br>Q<br>Q | 99<br>287, 288<br>287, 289<br>287, 290<br>287, 291 |



Table A6.4: Oxygenated chlorocarbons (C, H, O, Cl) (. . . continued)

| Substance<br>Formula<br>(Trivial Name)<br>[CAS Registry Number]<br>InChIKey | $H_s^{cp}$<br>(at $T^\ominus$)<br>$\left[\dfrac{\mathrm{mol}}{\mathrm{m^3\,Pa}}\right]$ | $\dfrac{\mathrm{d}\ln H_s^{cp}}{\mathrm{d}(1/T)}$<br><br>[K] | Reference | Type | Note |
|---|---|---|---|---|---|
| 2,4'-dichlorobenzophenone | 9.2 | | Zhang et al. (2010) | Q | 287, 288 |
| $C_{13}H_8Cl_2O$ | 6.9 | | Zhang et al. (2010) | Q | 287, 289 |
| [85-29-0] | $4.3\times10^1$ | | Zhang et al. (2010) | Q | 287, 290 |
| YXMYPHLWXBXNFF-UHFFFAOYSA-N | $6.1\times10^1$ | | Zhang et al. (2010) | Q | 287, 291 |
| 1-(4-chlorophenyl)-4,4- | 4.8 | | Zhang et al. (2010) | Q | 287, 288 |
| dimethylpent-1-en-3-one | | | | | |
| $C_{13}H_{15}ClO$ | 2.2 | | Zhang et al. (2010) | Q | 287, 289 |
| [1577-03-3] | 8.2 | | Zhang et al. (2010) | Q | 287, 290 |
| LXJZYHPGRKBVGF-UHFFFAOYSA-N | 1.5 | | Zhang et al. (2010) | Q | 287, 291 |
| dichlorophen | $8.2\times10^6$ | | HSDB (2015) | V | |
| $C_{13}H_{10}Cl_2O_2$ | $8.5\times10^6$ | | Mackay et al. (2006d) | V | |
| [97-23-4] | | | | | |
| MDNWOSOZYLHTCG-UHFFFAOYSA-N | | | | | |
| MCPB ethyl ester | $2.0\times10^2$ | | Duchowicz et al. (2020) | V | 186 |
| $C_{13}H_{17}ClO_3$ | 2.9 | | Duchowicz et al. (2020) | Q | |
| [10443-70-6] | | | | | |
| XNKARWLGLZGMGX-UHFFFAOYSA-N | | | | | |
| 2-chloro-9,10-anthracenedione | $4.2\times10^3$ | | Zhang et al. (2010) | Q | 287, 288 |
| $C_{14}H_7ClO_2$ | $6.7\times10^2$ | | Zhang et al. (2010) | Q | 287, 289 |
| [131-09-9] | $1.4\times10^2$ | | Zhang et al. (2010) | Q | 287, 290 |
| FPKCTSIVDAWGFA-UHFFFAOYSA-N | $3.3\times10^4$ | | Zhang et al. (2010) | Q | 287, 291 |
| dicofol | $4.1\times10^1$ | | Duchowicz et al. (2020) | V | 186 |
| $C_{14}H_9Cl_5O$ | $4.1\times10^1$ | | HSDB (2015) | V | |
| [115-32-2] | $9.6\times10^2$ | | Duchowicz et al. (2020) | Q | |
| UOAMTSKGCBMZTC-UHFFFAOYSA-N | $1.8\times10^4$ | | Zhang et al. (2010) | Q | 287, 288 |
| | $3.1\times10^2$ | | Zhang et al. (2010) | Q | 287, 289 |
| | $9.2\times10^1$ | | Zhang et al. (2010) | Q | 287, 290 |
| | $3.2\times10^3$ | | Zhang et al. (2010) | Q | 287, 291 |
| bis(2,4-dichlorobenzoyl)peroxide | 9.2 | | Zhang et al. (2010) | Q | 287, 288 |
| $C_{14}H_6Cl_4O_4$ | $1.4\times10^2$ | | Zhang et al. (2010) | Q | 287, 289 |
| [133-14-2] | $1.6\times10^3$ | | Zhang et al. (2010) | Q | 287, 290 |
| WRXCBRHBHGNNQA-UHFFFAOYSA-N | $3.5\times10^3$ | | Zhang et al. (2010) | Q | 287, 291 |
| dipropyl tetrachlorophthalate | $4.7\times10^1$ | | Zhang et al. (2010) | Q | 287, 288 |
| $C_{14}H_{14}Cl_4O_4$ | $3.0\times10^1$ | | Zhang et al. (2010) | Q | 287, 289 |
| [6928-67-2] | $1.0\times10^1$ | | Zhang et al. (2010) | Q | 287, 290 |
| QJRSPJSHRYYBJL-UHFFFAOYSA-N | $2.8\times10^1$ | | Zhang et al. (2010) | Q | 287, 291 |
| 2-(4-chlorobenzoyl)benzoic acid | $3.4\times10^5$ | | Zhang et al. (2010) | Q | 287, 288 |
| $C_{14}H_9ClO_3$ | $3.6\times10^4$ | | Zhang et al. (2010) | Q | 287, 289 |
| [85-56-3] | $7.9\times10^7$ | | Zhang et al. (2010) | Q | 287, 290 |
| YWECCEXWKFHHQJ-UHFFFAOYSA-N | $2.1\times10^6$ | | Zhang et al. (2010) | Q | 287, 291 |



Table A6.4: Oxygenated chlorocarbons (C, H, O, Cl) (... continued)

| Substance Formula (Trivial Name) [CAS Registry Number] InChIKey | $H_s^{cp}$ (at $T^{\ominus}$) $\left[\dfrac{\mathrm{mol}}{\mathrm{m^3\,Pa}}\right]$ | $\dfrac{\mathrm{d}\ln H_s^{cp}}{\mathrm{d}(1/T)}$ [K] | Reference | Type | Note |
|---|---|---|---|---|---|
| (2,4,5-trichlorophenoxy)acetic acid butoxyethanol ester $C_{14}H_{17}Cl_3O_4$ [2545-59-7] GLDWASBMYWLQGG-UHFFFAOYSA-N | $1.2\times10^2$ | | HSDB (2015) | Q | 99 |
| (2,4-dichlorophenoxy)-acetic acid, 2-butoxyethyl ester $C_{14}H_{18}Cl_2O_4$ [1929-73-3] ZMWGIGHRZQTQRE-UHFFFAOYSA-N | $6.2\times10^1$ $6.2\times10^1$ $6.0\times10^1$ | | Duchowicz et al. (2020) HSDB (2015) Duchowicz et al. (2020) | V V Q | 186 |
| 1-chloro-9,10-anthracenedione $C_{14}H_7ClO_2$ (1-chloroanthraquinone) [82-44-0] BOCJQSFSGAZAPQ-UHFFFAOYSA-N | $4.3\times10^2$ $4.2\times10^3$ | | Abraham et al. (2019) HSDB (2015) | Q Q | 99 |
| chlorflurenol methyl $C_{15}H_{11}ClO_3$ [2536-31-4] LINPVWIEWJTEEJ-UHFFFAOYSA-N | $1.3\times10^3$ | | Ebert et al. (2023) | ? | 365 |
| 4,4'-(1-methylethylidene)bis(2,6-dichlorophenol) $C_{15}H_{12}Cl_4O_2$ (2,2',6,6'-tetrachlorobisphenol A) [79-95-8] KYPYTERUKNKOLP-UHFFFAOYSA-N | $3.5\times10^6$ | | HSDB (2015) | Q | 447 |
| diclofop $C_{15}H_{12}Cl_2O_4$ [40843-25-2] OOLBCHYXZDXLDS-UHFFFAOYSA-N | $9.2\times10^5$ | | Ebert et al. (2023) | ? | 316 |
| methoxychlor $C_{16}H_{15}Cl_3O_2$ [72-43-5] IAKOZHOLGAGEJT-UHFFFAOYSA-N | $4.9\times10^1$ $1.0$ $4.0$ $5.5$ $2.8$ $6.4$ $4.9\times10^1$ | | Altschuh et al. (1999) Mackay et al. (2006d) Keshavarz et al. (2022) Duchowicz et al. (2020) Hilal et al. (2008) Modarresi et al. (2007) Duchowicz et al. (2020) | M V Q Q Q Q ? | 67 185, 21 |
| diclofop-methyl $C_{16}H_{14}Cl_2O_4$ [51338-27-3] BACHBFVBHLGWSL-UHFFFAOYSA-N | $5.0$ $5.0$ $1.3\times10^2$ $2.6\times10^2$ $9.5\times10^1$ | | Duchowicz et al. (2020) Mackay et al. (2006d) Duchowicz et al. (2020) HSDB (2015) Maniere et al. (2011) | V V Q Q ? | 186 99 12, 165 |



Table A6.4: Oxygenated chlorocarbons (C, H, O, Cl) (...continued)

| Substance Formula (Trivial Name) [CAS Registry Number] InChIKey | $H_s^{cp}$ (at $T^\ominus$) $\left[\dfrac{\mathrm{mol}}{\mathrm{m^3\,Pa}}\right]$ | $\dfrac{\mathrm{d}\ln H_s^{cp}}{\mathrm{d}(1/T)}$ [K] | Reference | Type | Note |
|---|---|---|---|---|---|
| chlorobenzilate $C_{16}H_{14}Cl_2O_3$ [510-15-6] RAPBNVDSDCTNRC-UHFFFAOYSA-N | $1.4\times10^2$ $2.6\times10^2$ | | HSDB (2015) MacBean (2012a) | V ? | |
| (2,4-dichlorophenoxy)-acetic acid 2-ethylhexyl ester $C_{16}H_{22}Cl_2O_3$ [1928-43-4] QZSFJRIWRPJUOH-UHFFFAOYSA-N | $5.5\times10^{-1}$ $5.5\times10^{-1}$ | | MacBean (2012b) Maniere et al. (2011) | X ? | 350 241, 165 |
| (2,4-dichlorophenoxy)-acetic acid, isooctyl ester $C_{16}H_{22}Cl_2O_3$ [25168-26-7] BBPLSOGERZQYQC-UHFFFAOYSA-N | $1.0\times10^{-1}$ $3.3\times10^{-1}$ $1.7\times10^{-1}$ | | Duchowicz et al. (2020) Duchowicz et al. (2020) HSDB (2015) | V Q Q | 186 99 |
| mecoprop-2-butoxyethyl ester $C_{16}H_{23}ClO_4$ [23359-62-8] GWFGUAFFJVHZKY-UHFFFAOYSA-N | 5.0 | | Ebert et al. (2023) | ? | 318 |
| chloropropylate $C_{17}H_{16}Cl_2O_3$ [5836-10-2] AXGUBXVWZBFQGA-UHFFFAOYSA-N | $1.2\times10^3$ $1.0\times10^2$ | | HSDB (2015) MacBean (2012a) | V ? | |
| MCPA-2-ethylhexyl ester $C_{17}H_{25}ClO_3$ [29450-45-1] IDGRPSMONFWWEK-UHFFFAOYSA-N | $2.3\times10^{-1}$ | | Ebert et al. (2023) | ? | 318 |
| 1-(2-(2-chloroethoxy)ethoxy)-4-(1,1,3,3-tetramethylbutyl)benzene $C_{18}H_{29}ClO_2$ [65925-28-2] FITQCDWGUKECBJ-UHFFFAOYSA-N | $5.3\times10^{-1}$ 1.6 $3.8\times10^{-1}$ $8.4\times10^{-2}$ | | Zhang et al. (2010) Zhang et al. (2010) Zhang et al. (2010) Zhang et al. (2010) | Q Q Q Q | 287, 288 287, 289 287, 290 287, 291 |
| fenofibrate $C_{20}H_{21}ClO_4$ [49562-28-9] YMTINGFKWWXKFG-UHFFFAOYSA-N | $2.2\times10^3$ | | HSDB (2015) | Q | 99 |
| indanofan $C_{20}H_{17}O_3Cl$ [133220-30-1] PMAAYIYCDXGUAP-UHFFFAOYSA-N | $1.5\times10^4$ $5.4\times10^3$ | | Duchowicz et al. (2020) Duchowicz et al. (2020) | V Q | 186 |



Table A6.4: Oxygenated chlorocarbons (C, H, O, Cl) (. . . continued)

| Substance / Formula / (Trivial Name) / [CAS Registry Number] / InChIKey | $H_s^{cp}$ (at $T^\ominus$) $\left[\dfrac{\mathrm{mol}}{\mathrm{m^3\,Pa}}\right]$ | $\dfrac{\mathrm{d}\ln H_s^{cp}}{\mathrm{d}(1/T)}$ [K] | Reference | Type | Note |
|---|---|---|---|---|---|
| spirodiclofen $C_{21}H_{24}Cl_2O_4$ [148477-71-8] DTDSAWVUFPGDMX-UHFFFAOYSA-N | $1.7\times10^2$ $>5.0\times10^2$ | | HSDB (2015) Maniere et al. (2011) | V ? | 12, 165 |
| permethrin $C_{21}H_{20}Cl_2O_3$ [52645-53-1] RLLPVAHGXHCWKJ-UHFFFAOYSA-N | 5.3 4.1 9.0 $1.2\times10^1$ | | Duchowicz et al. (2020) HSDB (2015) Mackay et al. (2006d) Duchowicz et al. (2020) | V V V Q | 186 |
| chlorophacinone $C_{23}H_{15}ClO_3$ [3691-35-8] UDHXJZHVNHGCEC-UHFFFAOYSA-N | $2.7\times10^6$ $4.1\times10^4$ | | Duchowicz et al. (2020) Duchowicz et al. (2020) | V Q | 186 |
| chlormadinone acetate $C_{23}H_{29}ClO_4$ [302-22-7] QMBJSIBWORFWQT-UHFFFAOYSA-N | $1.8\times10^4$ | | HSDB (2015) | Q | 99 |
| 3,4,5,6-tetrachlorophthalic acid bis(2-ethylhexyl) ester $C_{24}H_{34}Cl_4O_4$ [34832-88-7] BELGUQVGZMFEAQ-UHFFFAOYSA-N | 2.8 $2.3\times10^1$ $1.0\times10^4$ 3.5 | | Zhang et al. (2010) Zhang et al. (2010) Zhang et al. (2010) Zhang et al. (2010) | Q Q Q Q | 287, 288 287, 289 287, 290 287, 291 |
| endosulfan alcohol $C_9H_8Cl_6O_2$ [2157-19-9] GTSJHTSVFKEASK-UHFFFAOYSA-N | $7.7\times10^3$ $3.0\times10^6$ $1.3\times10^5$ $1.8\times10^5$ | | Zhang et al. (2010) Zhang et al. (2010) Zhang et al. (2010) Zhang et al. (2010) | Q Q Q Q | 287, 288 287, 289 287, 290 287, 291 |
| chlorendic anhydride $C_9H_2Cl_6O_3$ [115-27-5] FLBJFXNAEMSXGL-UHFFFAOYSA-N | $1.1\times10^2$ $3.1\times10^4$ $1.5\times10^4$ $3.9\times10^7$ | | Zhang et al. (2010) Zhang et al. (2010) Zhang et al. (2010) Zhang et al. (2010) | Q Q Q Q | 287, 288 287, 289 287, 290 287, 291 |
| 1,4,5,6,7,7-hexachloro-bicyclo[2.2.1]hept-5-ene-2,3-dicarboxylic acid $C_9H_4Cl_6O_4$ [115-28-6] DJKGDNKYTKCJKD-UHFFFAOYSA-N | $3.3\times10^8$ $3.3\times10^8$ $3.1\times10^9$ $3.9\times10^9$ $7.3\times10^7$ | | HSDB (2015) Zhang et al. (2010) Zhang et al. (2010) Zhang et al. (2010) Zhang et al. (2010) | Q Q Q Q Q | 99 287, 288 287, 289 287, 290 287, 291 |



Table A6.4: Oxygenated chlorocarbons (C, H, O, Cl) (...continued)

| Substance<br>Formula<br>(Trivial Name)<br>[CAS Registry Number]<br>InChIKey | $H_s^{cp}$<br>(at $T^\ominus$)<br>$\left[\dfrac{\mathrm{mol}}{\mathrm{m^3\,Pa}}\right]$ | $\dfrac{\mathrm{d}\ln H_s^{cp}}{\mathrm{d}(1/T)}$<br><br>[K] | Reference | Type | Note |
|---|---|---|---|---|---|
| heptachlorepoxide | $4.8\times10^{-1}$ | | Shen and Wania (2005) | L | 366 |
| $C_{10}H_5Cl_7O$ | $5.9\times10^{-1}$ | | Shen and Wania (2005) | L | 367 |
| [1024-57-3] | $9.6\times10^{-1}$ | | Chao et al. (2017) | M | |
| ZXFXBSWRVIQKOD-UHFFFAOYSA-N | $5.0\times10^{-1}$ | 5200 | Cetin et al. (2006) | M | |
| | $4.7\times10^{-1}$ | | Altschuh et al. (1999) | M | |
| | $3.1\times10^{-1}$ | | Warner et al. (1980) | M | |
| | $5.4\times10^{-1}$ | | Hilal et al. (2008) | C | |
| | $1.3\times10^{1}$ | | Ryan et al. (1988) | C | |
| | $3.1\times10^{-1}$ | | Shen (1982) | C | |
| | $6.6\times10^{-1}$ | | Keshavarz et al. (2022) | Q | |
| | $7.7\times10^{-1}$ | | Duchowicz et al. (2020) | Q | |
| | $7.3$ | | Hilal et al. (2008) | Q | |
| | $5.4$ | | Modarresi et al. (2007) | Q | 67 |
| | $4.7\times10^{-1}$ | | Duchowicz et al. (2020) | ? | 185, 21 |
| dieldrin | $1.0$ | | Shen and Wania (2005) | L | 366 |
| $C_{12}H_8OCl_6$ | $9.1\times10^{-1}$ | | Shen and Wania (2005) | L | 367 |
| [60-57-1] | $9.1\times10^{-1}$ | | Mackay and Shiu (1981) | L | |
| DFBKLUNHFCTMDC-NLUYNBKHSA-N | $6.5\times10^{-1}$ | | Chao et al. (2017) | M | |
| | $9.2\times10^{-1}$ | 5800 | Cetin et al. (2006) | M | |
| | $9.8\times10^{-1}$ | | Altschuh et al. (1999) | M | |
| | $3.4\times10^{-1}$ | | Slater and Spedding (1981) | M | 12 |
| | $1.7\times10^{-1}$ | | Warner et al. (1980) | M | |
| | $8.9\times10^{-1}$ | | Mackay et al. (2006d) | V | |
| | $8.9\times10^{-1}$ | | Suntio et al. (1988) | V | 12 |
| | $4.9\times10^{1}$ | | Mackay and Leinonen (1975) | V | |
| | $8.8\times10^{-3}$ | | Barcelo and Hennion (1997) | X | 567 |
| | $1.7\times10^{-1}$ | | Hilal et al. (2008) | C | |
| | $5.0\times10^{1}$ | | Suntio et al. (1988) | C | 12 |
| | $2.2\times10^{1}$ | | Suntio et al. (1988) | C | 681 |
| | $9.8\times10^{-1}$ | | Suntio et al. (1988) | C | 681 |
| | $1.7\times10^{-1}$ | | Suntio et al. (1988) | C | |
| | $4.7\times10^{-2}$ | | Suntio et al. (1988) | C | |
| | $1.3$ | | Ryan et al. (1988) | C | |
| | $1.7\times10^{-1}$ | | Shen (1982) | C | |
| | $2.4\times10^{-1}$ | | Keshavarz et al. (2022) | Q | |
| | $7.2\times10^{-1}$ | | Duchowicz et al. (2020) | Q | |
| | $7.4\times10^{-4}$ | | Goodarzi et al. (2010) | Q | 568, 571 |
| | $1.1$ | | Hilal et al. (2008) | Q | |
| | $2.7$ | | Modarresi et al. (2007) | Q | 67 |
| | $9.9\times10^{-1}$ | | Duchowicz et al. (2020) | ? | 185, 21 |
| | $1.2$ | | MacBean (2012a) | ? | |
| | $5.7\times10^{1}$ | | Brimblecombe (1986) | ? | 80 |



Table A6.4: Oxygenated chlorocarbons (C, H, O, Cl) (...continued)

| Substance Formula (Trivial Name) [CAS Registry Number] InChIKey | $H_s^{cp}$ (at $T^{\ominus}$) $\left[\dfrac{\mathrm{mol}}{\mathrm{m^3\,Pa}}\right]$ | $\dfrac{\mathrm{d}\ln H_s^{cp}}{\mathrm{d}(1/T)}$ [K] | Reference | Type | Note |
|---|---|---|---|---|---|
| endrin | 1.6 | | Shen and Wania (2005) | L | 366 |
| $C_{12}H_8Cl_6O$ | $9.1\times10^{-1}$ | | Shen and Wania (2005) | L | 367 |
| [72-20-8] | 1.5 | | Chao et al. (2017) | M | |
| DFBKLUNHFCTMDC-GKRDHZSOSA-N | 1.8 | 4600 | Cetin et al. (2006) | M | |
| | 1.6 | | Altschuh et al. (1999) | M | |
| | $3.0\times10^1$ | | Mackay et al. (2006d) | V | |
| | $3.0\times10^1$ | | Suntio et al. (1988) | V | 12 |
| | $5.6\times10^3$ | | Suntio et al. (1988) | C | |
| | $2.4\times10^1$ | | Ryan et al. (1988) | C | |
| | 1.2 | | Keshavarz et al. (2022) | Q | |
| | $7.2\times10^{-1}$ | | Duchowicz et al. (2020) | Q | 184 |
| | 1.1 | | Hilal et al. (2008) | Q | |
| | 2.7 | | Modarresi et al. (2007) | Q | 67 |
| | 1.6 | | Duchowicz et al. (2020) | ? | 185, 21 |
| 1,4,5,6,7,7-hexachlorobicyclo[2.2.1]hept-5-ene-2,3-dicarboxylic acid, dibutyl ester | $5.8\times10^2$ | | Zhang et al. (2010) | Q | 287, 288 |
| $C_{17}H_{20}Cl_6O_4$ | $1.4\times10^2$ | | Zhang et al. (2010) | Q | 287, 289 |
| [1770-80-5] | $4.6\times10^3$ | | Zhang et al. (2010) | Q | 287, 290 |
| UJAHPBDUQZFDLA-UHFFFAOYSA-N | $8.0\times10^2$ | | Zhang et al. (2010) | Q | 287, 291 |
| di-2-ethylhexyl chlorendate | $6.0\times10^1$ | | Zhang et al. (2010) | Q | 287, 288 |
| $C_{25}H_{36}Cl_6O_4$ | $2.1\times10^2$ | | Zhang et al. (2010) | Q | 287, 289 |
| [4827-55-8] | $5.2\times10^3$ | | Zhang et al. (2010) | Q | 287, 290 |
| ONIHOIFWLALAQH-UHFFFAOYSA-N | $1.6\times10^2$ | | Zhang et al. (2010) | Q | 287, 291 |



### A6.5 Polychlorinated diphenyl ethers (PCDEs)

Table A6.5: Polychlorinated diphenyl ethers (PCDEs)

| Substance<br>Formula<br>(Trivial Name)<br>[CAS Registry Number]<br>InChIKey | $H_s^{cp}$ (at $T^\ominus$) $\left[\dfrac{\mathrm{mol}}{\mathrm{m}^3\,\mathrm{Pa}}\right]$ | $\dfrac{\mathrm{d}\ln H_s^{cp}}{\mathrm{d}(1/T)}$ [K] | Reference | Type | Note |
|---|---|---|---|---|---|
| 2-chlorodiphenyl ether<br>$C_{12}H_9ClO$<br>(PCDE-1)<br>[2689-07-8]<br>IPBRZLMGGXHHMS-UHFFFAOYSA-N | $3.1\times10^{-2}$ | | Kurz and Ballschmiter (1999) | V | |
| 3-chlorodiphenyl ether<br>$C_{12}H_9ClO$<br>(PCDE-2)<br>[6452-49-9]<br>BMURONZFJJPAOK-UHFFFAOYSA-N | $1.2\times10^{-1}$<br>$2.7\times10^{-2}$<br>$3.0\times10^{-1}$ | | Kurz and Ballschmiter (1999)<br>Hilal et al. (2008)<br>Modarresi et al. (2007) | V<br>Q<br>Q | <br><br>67 |
| 4-chlorodiphenyl ether<br>$C_{12}H_9ClO$<br>(PCDE-3)<br>[7005-72-3]<br>PGPNJCAMHOJTEF-UHFFFAOYSA-N | $1.1\times10^{-1}$<br>$4.5\times10^{-2}$<br>$9.0\times10^{-2}$<br>$4.0\times10^{-2}$<br>$3.1\times10^{-2}$ | | Kurz and Ballschmiter (1999)<br>Mackay et al. (1993)<br>Howard and Meylan (1997)<br>Ryan et al. (1988)<br>Hilal et al. (2008) | V<br>V<br>X<br>C<br>Q | <br><br>446<br><br> |
| 2,3-dichlorodiphenyl ether<br>$C_{12}H_8Cl_2O$<br>(PCDE-5)<br>[68486-28-2]<br>VSKSUBSGORDMQX-UHFFFAOYSA-N | $2.4\times10^{-1}$ | | Kurz and Ballschmiter (1999) | V | |
| 2,4-dichlorodiphenyl ether<br>$C_{12}H_8Cl_2O$<br>(PCDE-7)<br>[51892-26-3]<br>KXIPYLZZJZMMPD-UHFFFAOYSA-N | $1.9\times10^{-1}$ | | Kurz and Ballschmiter (1999) | V | |
| 2,4'-dichlorodiphenyl ether<br>$C_{12}H_8Cl_2O$<br>(PCDE-8)<br>[6903-65-7]<br>MWKULKMSBBSGTP-UHFFFAOYSA-N | $3.2\times10^{-2}$ | | Kurz and Ballschmiter (1999) | V | |
| 2,5-dichlorodiphenyl ether<br>$C_{12}H_8Cl_2O$<br>(PCDE-9)<br>[24910-69-8]<br>VITXVDNQHXYQSK-UHFFFAOYSA-N | $7.9\times10^{-2}$ | | Kurz and Ballschmiter (1999) | V | |
| 2,6-dichlorodiphenyl ether<br>$C_{12}H_8Cl_2O$<br>(PCDE-10)<br>[28419-69-4]<br>IRLZOQDGEAEIPX-UHFFFAOYSA-N | $5.0\times10^{-2}$ | | Kurz and Ballschmiter (1999) | V | |



Table A6.5: Polychlorinated diphenyl ethers (PCDEs) (... continued)

| Substance Formula (Trivial Name) [CAS Registry Number] InChIKey | $H_s^{cp}$ (at $T^{\ominus}$) $\left[\dfrac{\text{mol}}{\text{m}^3\,\text{Pa}}\right]$ | $\dfrac{\text{d}\ln H_s^{cp}}{\text{d}(1/T)}$ [K] | Reference | Type | Note |
|---|---|---|---|---|---|
| 3,4-dichlorodiphenyl ether $C_{12}H_8Cl_2O$ (PCDE-12) [55538-69-7] QFQLZVAAQOJUFS-UHFFFAOYSA-N | $1.1\times10^{-1}$ | | Kurz and Ballschmiter (1999) | V | |
| 3,4'-dichlorodiphenyl ether $C_{12}H_8Cl_2O$ (PCDE-13) [6842-62-2] HPRGYUWRGCTBAV-UHFFFAOYSA-N | $1.3\times10^{-1}$ | | Kurz and Ballschmiter (1999) | V | |
| 3,5-dichlorodiphenyl ether $C_{12}H_8Cl_2O$ (PCDE-14) [24910-68-7] LEKOWSSKXYFERR-UHFFFAOYSA-N | $6.5\times10^{-2}$ | | Kurz and Ballschmiter (1999) | V | |
| 4,4'-dichlorodiphenyl ether $C_{12}H_8Cl_2O$ (PCDE-15) [2444-89-5] URUJZHZLCCIILC-UHFFFAOYSA-N | $2.1\times10^{-1}$ | | Kurz and Ballschmiter (1999) | V | |
| 2,2',4-trichlorodiphenyl ether $C_{12}H_7Cl_3O$ (PCDE-17) [68914-97-6] YXMNUPKWUMIZAV-UHFFFAOYSA-N | $4.5\times10^{-1}$ | | Kurz and Ballschmiter (1999) | V | |
| 2,3,4-trichlorodiphenyl ether $C_{12}H_7Cl_3O$ (PCDE-21) [85918-32-7] ANKBTLMYMVMWBS-UHFFFAOYSA-N | $2.8\times10^{-1}$ | | Kurz and Ballschmiter (1999) | V | |
| 2,3,4'-trichlorodiphenyl ether $C_{12}H_7Cl_3O$ (PCDE-22) [157683-71-1] KOTNFWJIMRGVBR-UHFFFAOYSA-N | $3.2\times10^{-1}$ | | Kurz and Ballschmiter (1999) | V | |
| 2,3,5-trichlorodiphenyl ether $C_{12}H_7Cl_3O$ (PCDE-23) [162853-24-9] MDSPKCWGOVVFJD-UHFFFAOYSA-N | $2.2\times10^{-1}$ | | Kurz and Ballschmiter (1999) | V | |



Table A6.5: Polychlorinated diphenyl ethers (PCDEs) (... continued)

| Substance Formula (Trivial Name) [CAS Registry Number] InChIKey | $H_s^{cp}$ (at $T^\ominus$) $\left[\dfrac{\mathrm{mol}}{\mathrm{m^3\,Pa}}\right]$ | $\dfrac{\mathrm{d}\ln H_s^{cp}}{\mathrm{d}(1/T)}$ [K] | Reference | Type | Note |
|---|---|---|---|---|---|
| 2,3,6-trichlorodiphenyl ether $C_{12}H_7Cl_3O$ (PCDE-24) [162853-25-0] RQSRPDGSUDRYLH-UHFFFAOYSA-N | $3.0\times10^{-2}$ | | Kurz and Ballschmiter (1999) | V | |
| 2,3',4-trichlorodiphenyl ether $C_{12}H_7Cl_3O$ (PCDE-25) [155999-93-2] BJFOCMRBCBZFJT-UHFFFAOYSA-N | $1.5\times10^{-1}$ | | Kurz and Ballschmiter (1999) | V | |
| 2,4,4'-trichlorodiphenyl ether $C_{12}H_7Cl_3O$ (PCDE-28) [59039-21-3] PIORTDHJOLELKR-UHFFFAOYSA-N | $3.0\times10^{-2}$ | | Kurz and Ballschmiter (1999) | V | |
| 2,4,5-trichlorodiphenyl ether $C_{12}H_7Cl_3O$ (PCDE-29) [52322-80-2] UWKZWXCTDPYXHU-UHFFFAOYSA-N | $8.9\times10^{-3}$ | | Kurz and Ballschmiter (1999) | V | |
| 2,4,6-trichlorodiphenyl ether $C_{12}H_7Cl_3O$ (PCDE-30) [63646-52-6] WXLQUFLKMICSSY-UHFFFAOYSA-N | $1.4\times10^{-2}$ | | Kurz and Ballschmiter (1999) | V | |
| 2,4',5-trichlorodiphenyl ether $C_{12}H_7Cl_3O$ (PCDE-31) [65075-00-5] FZBSTAVCYOMFMD-UHFFFAOYSA-N | $1.6\times10^{-1}$ | | Kurz and Ballschmiter (1999) | V | |
| 2,4',6-trichlorodiphenyl ether $C_{12}H_7Cl_3O$ (PCDE-32) [157683-72-2] ZMRFCSWFCKICQE-UHFFFAOYSA-N | $4.2\times10^{-2}$ | | Kurz and Ballschmiter (1999) | V | |
| 2,3',4'-trichlorodiphenyl ether $C_{12}H_7Cl_3O$ (PCDE-33) [61328-44-7] VBNJGYOQZAENPO-UHFFFAOYSA-N | $3.4\times10^{-1}$ | | Kurz and Ballschmiter (1999) | V | |



Table A6.5: Polychlorinated diphenyl ethers (PCDEs) (... continued)

| Substance Formula (Trivial Name) [CAS Registry Number] InChIKey | $H_s^{cp}$ (at $T^{\ominus}$) $\left[\dfrac{\text{mol}}{\text{m}^3\,\text{Pa}}\right]$ | $\dfrac{\text{d}\ln H_s^{cp}}{\text{d}(1/T)}$ [K] | Reference | Type | Note |
|---|---|---|---|---|---|
| 3,3',4-trichlorodiphenyl ether $C_{12}H_7Cl_3O$ (PCDE-35) [66794-60-3] BUKLLCIUMZELOH-UHFFFAOYSA-N | $2.2\times10^{-1}$ | | Kurz and Ballschmiter (1999) | V | |
| 3,4,4'-trichlorodiphenyl ether $C_{12}H_7Cl_3O$ (PCDE-37) [63646-51-5] FTZDEZOARSJHGU-UHFFFAOYSA-N | $1.6\times10^{-1}$ | | Kurz and Ballschmiter (1999) | V | |
| 3,4,5-trichlorodiphenyl ether $C_{12}H_7Cl_3O$ (PCDE-38) [63646-53-7] ZQCCHNBMSHOJAT-UHFFFAOYSA-N | $9.1\times10^{-3}$ | | Kurz and Ballschmiter (1999) | V | |
| 3,4',5-trichlorodiphenyl ether $C_{12}H_7Cl_3O$ (PCDE-39) [24910-73-4] CCEZFZGXWFJQEF-UHFFFAOYSA-N | $1.6\times10^{-1}$ | | Kurz and Ballschmiter (1999) | V | |
| 2,2',3,4-tetrachlorodiphenyl ether $C_{12}H_6Cl_4O$ (PCDE-41) [220002-37-9] VTTDWGUAWMLHPR-UHFFFAOYSA-N | $5.5\times10^{-2}$ | | Kurz and Ballschmiter (1999) | V | |
| 2,2',3,4'-tetrachlorodiphenyl ether $C_{12}H_6Cl_4O$ (PCDE-42) [147102-63-4] UYVBJPIPRAJEKW-UHFFFAOYSA-N | $5.8\times10^{-2}$ | | Kurz and Ballschmiter (1999) | V | |
| 2,2',4,4'-tetrachlorodiphenyl ether $C_{12}H_6Cl_4O$ (PCDE-47) [28076-73-5] ZRWRPGGXCSSBAO-UHFFFAOYSA-N | $2.9\times10^{-2}$ $2.8\times10^{-1}$ | | Kurz and Ballschmiter (1999) HSDB (2015) | V Q | 99 |
| 2,2',4,5-tetrachlorodiphenyl ether $C_{12}H_6Cl_4O$ (PCDE-48) [162853-26-1] DRVAMLZVBNYSQE-UHFFFAOYSA-N | $1.6\times10^{-2}$ | | Kurz and Ballschmiter (1999) | V | |



Table A6.5: Polychlorinated diphenyl ethers (PCDEs) (...continued)

| Substance<br>Formula<br>(Trivial Name)<br>[CAS Registry Number]<br>InChIKey | $H_s^{cp}$<br>(at $T^{\ominus}$)<br><br>$\left[\dfrac{\text{mol}}{\text{m}^3\,\text{Pa}}\right]$ | $\dfrac{\mathrm{d}\ln H_s^{cp}}{\mathrm{d}(1/T)}$<br><br><br>[K] | Reference | Type | Note |
|---|---|---|---|---|---|
| 2,2',4,5'-tetrachlorodiphenyl ether<br>$C_{12}H_6Cl_4O$<br>(PCDE-49)<br>[155999-92-1]<br>XIXPEFLLHQAIOY-UHFFFAOYSA-N | $2.6\times10^{-2}$ | | Kurz and Ballschmiter (1999) | V | |
| 2,3,3',4-tetrachlorodiphenyl ether<br>$C_{12}H_6Cl_4O$<br>(PCDE-55)<br>[220002-39-1]<br>IZISEKJPNSBPRX-UHFFFAOYSA-N | $2.3\times10^{-2}$ | | Kurz and Ballschmiter (1999) | V | |
| 2,3,3',4'-tetrachlorodiphenyl ether<br>$C_{12}H_6Cl_4O$<br>(PCDE-56)<br>[162853-27-2]<br>JVFGXWCLGNACOG-UHFFFAOYSA-N | $4.4\times10^{-2}$ | | Kurz and Ballschmiter (1999) | V | |
| 2,3,4,4'-tetrachlorodiphenyl ether<br>$C_{12}H_6Cl_4O$<br>(PCDE-60)<br>[65075-01-6]<br>ZPOBHUZRVVTGKB-UHFFFAOYSA-N | $3.6\times10^{-2}$ | | Kurz and Ballschmiter (1999) | V | |
| 2,3,4,5-tetrachlorodiphenyl ether<br>$C_{12}H_6Cl_4O$<br>(PCDE-61)<br>[220002-40-4]<br>QAPPZLVGIXQUJW-UHFFFAOYSA-N | $6.6\times10^{-3}$ | | Kurz and Ballschmiter (1999) | V | |
| 2,3,4,6-tetrachlorodiphenyl ether<br>$C_{12}H_6Cl_4O$<br>(PCDE-62)<br>[85918-33-8]<br>WJZPRRFFZWVZJL-UHFFFAOYSA-N | $9.1\times10^{-3}$ | | Kurz and Ballschmiter (1999) | V | |
| 2,3,4',5-tetrachlorodiphenyl ether<br>$C_{12}H_6Cl_4O$<br>(PCDE-63)<br>[220002-41-5]<br>YHQRYZGDQCSHFO-UHFFFAOYSA-N | $1.5\times10^{-2}$ | | Kurz and Ballschmiter (1999) | V | |
| 2,3,4',6-tetrachlorodiphenyl ether<br>$C_{12}H_6Cl_4O$<br>(PCDE-64)<br>[220002-42-6]<br>HJCNTJIMUWQMSA-UHFFFAOYSA-N | $3.8\times10^{-2}$ | | Kurz and Ballschmiter (1999) | V | |



Table A6.5: Polychlorinated diphenyl ethers (PCDEs) (...continued)

| Substance<br>Formula<br>(Trivial Name)<br>[CAS Registry Number]<br>InChIKey | $H_s^{cp}$<br>(at $T^{\ominus}$)<br>$\left[\dfrac{\text{mol}}{\text{m}^3\,\text{Pa}}\right]$ | $\dfrac{\text{d}\ln H_s^{cp}}{\text{d}(1/T)}$<br><br>[K] | Reference | Type | Note |
|---|---|---|---|---|---|
| 2,3,5,6-tetrachlorodiphenyl ether<br>$C_{12}H_6Cl_4O$<br>(PCDE-65)<br>[63646-54-8]<br>UFGXCDQXRUFDMA-UHFFFAOYSA-N | $9.3\times10^{-3}$ | | Kurz and Ballschmiter (1999) | V | |
| 2,3',4,4'-tetrachlorodiphenyl ether<br>$C_{12}H_6Cl_4O$<br>(PCDE-66)<br>[61328-46-9]<br>CAWSVYYOUZEMHA-UHFFFAOYSA-N | $2.5\times10^{-2}$ | | Kurz and Ballschmiter (1999) | V | |
| 2,3',4,5-tetrachlorodiphenyl ether<br>$C_{12}H_6Cl_4O$<br>(PCDE-67)<br>[152833-52-8]<br>QDQGECFQVPRMEO-UHFFFAOYSA-N | $8.9\times10^{-3}$ | | Kurz and Ballschmiter (1999) | V | |
| 2,3',4,5'-tetrachlorodiphenyl ether<br>$C_{12}H_6Cl_4O$<br>(PCDE-68)<br>[147102-64-5]<br>XORVNHCKWFPDLN-UHFFFAOYSA-N | $1.0\times10^{-2}$ | | Kurz and Ballschmiter (1999) | V | |
| 2,3',4',5-tetrachlorodiphenyl ether<br>$C_{12}H_6Cl_4O$<br>(PCDE-70)<br>[159553-67-0]<br>MFTPXFIGIJCNJM-UHFFFAOYSA-N | $1.8\times10^{-2}$ | | Kurz and Ballschmiter (1999) | V | |
| 2,3',4',6-tetrachlorodiphenyl ether<br>$C_{12}H_6Cl_4O$<br>(PCDE-71)<br>[130892-66-9]<br>IZZULMOBRTYHMH-UHFFFAOYSA-N | $4.6\times10^{-2}$ | | Kurz and Ballschmiter (1999) | V | |
| 2,4,4',5-tetrachlorodiphenyl ether<br>$C_{12}H_6Cl_4O$<br>(PCDE-74)<br>[61328-45-8]<br>KXOJXWNARBQHNX-UHFFFAOYSA-N | $1.9\times10^{-2}$ | | Kurz and Ballschmiter (1999) | V | |
| 2,4,4',6-tetrachlorodiphenyl ether<br>$C_{12}H_6Cl_4O$<br>(PCDE-75)<br>[63553-30-0]<br>YCWRSOTXXJYCKF-UHFFFAOYSA-N | $1.7\times10^{-2}$ | | Kurz and Ballschmiter (1999) | V | |



Table A6.5: Polychlorinated diphenyl ethers (PCDEs) (. . . continued)

| Substance Formula (Trivial Name) [CAS Registry Number] InChIKey | $H_s^{cp}$ (at $T^\ominus$) $\left[\dfrac{\text{mol}}{\text{m}^3\,\text{Pa}}\right]$ | $\dfrac{\text{d}\ln H_s^{cp}}{\text{d}(1/T)}$ [K] | Reference | Type | Note |
|---|---|---|---|---|---|
| 3,3',4,4'-tetrachlorodiphenyl ether $C_{12}H_6Cl_4O$ (PCDE-77) [56348-72-2] DHLVZXZRIZBPKG-UHFFFAOYSA-N | $4.1\times10^{-2}$ | | Kurz and Ballschmiter (1999) | V | |
| 3,3',4,5'-tetrachlorodiphenyl ether $C_{12}H_6Cl_4O$ (PCDE-79) [552884-22-7] BUPRSRBWDJVKGG-UHFFFAOYSA-N | $1.0\times10^{-2}$ | | Kurz and Ballschmiter (1999) | V | |
| 3,4,4',5-tetrachlorodiphenyl ether $C_{12}H_6Cl_4O$ (PCDE-81) [62615-07-0] CAUBBRCOHHHCNG-UHFFFAOYSA-N | $1.5\times10^{-2}$ | | Kurz and Ballschmiter (1999) | V | |
| 2,2',3,3',4-pentachlorodiphenyl ether $C_{12}H_5Cl_5O$ (PCDE-82) [160282-10-0] ABRDPORILGADAP-UHFFFAOYSA-N | $8.3\times10^{-2}$ | | Kurz and Ballschmiter (1999) | V | |
| 2,2',3,4,4'-pentachlorodiphenyl ether $C_{12}H_5Cl_5O$ (PCDE-85) [71585-37-0] RSBUDFTVQYJNHK-UHFFFAOYSA-N | $5.2\times10^{-2}$ | | Kurz and Ballschmiter (1999) | V | |
| 2,2',3,4,5'-pentachlorodiphenyl ether $C_{12}H_5Cl_5O$ (PCDE-87) [160282-09-7] SCLCXTQIVCULDR-UHFFFAOYSA-N | $2.2\times10^{-2}$ | | Kurz and Ballschmiter (1999) | V | |
| 2,2',3,4,6'-pentachlorodiphenyl ether $C_{12}H_5Cl_5O$ (PCDE-89) [85918-35-0] JGXAONSRHOSPTJ-UHFFFAOYSA-N | $6.5\times10^{-2}$ | | Kurz and Ballschmiter (1999) | V | |





Table A6.5: Polychlorinated diphenyl ethers (PCDEs) (. . . continued)

| Substance<br>Formula<br>(Trivial Name)<br>[CAS Registry Number]<br>InChIKey | $H_s^{cp}$<br>(at $T^{\ominus}$)<br>$\left[\dfrac{\text{mol}}{\text{m}^3\,\text{Pa}}\right]$ | $\dfrac{\text{d}\ln H_s^{cp}}{\text{d}(1/T)}$<br><br>[K] | Reference | Type | Note |
|---|---|---|---|---|---|
| 2,2',3,4',5-pentachlorodiphenyl ether<br>$C_{12}H_5Cl_5O$<br>(PCDE-90)<br>[157683-73-3]<br>QPEBRZQSEUOWPA-UHFFFAOYSA-N | $1.5\times10^{-2}$ | | Kurz and Ballschmiter (1999) | V | |
| 2,2',3,4',6-pentachlorodiphenyl ether<br>$C_{12}H_5Cl_5O$<br>(PCDE-91)<br>[116995-20-1]<br>BRNKJDZWZYZVNM-UHFFFAOYSA-N | $3.9\times10^{-2}$ | | Kurz and Ballschmiter (1999) | V | |
| 2,2',3,4',5'-pentachlorodiphenyl ether<br>$C_{12}H_5Cl_5O$<br>(PCDE-97)<br>[160282-08-6]<br>ABYZYVAIOYICJQ-UHFFFAOYSA-N | $3.3\times10^{-2}$ | | Kurz and Ballschmiter (1999) | V | |
| 2,2',4,4',5-pentachlorodiphenyl ether<br>$C_{12}H_5Cl_5O$<br>(PCDE-99)<br>[60123-64-0]<br>AVURWLKSFZBANQ-UHFFFAOYSA-N | $1.8\times10^{-2}$<br>$3.8\times10^{-2}$ | <br>6100 | Kurz and Ballschmiter (1999)<br>Paasivirta et al. (1999) | V<br>T | |
| 2,2',4,4',6-pentachlorodiphenyl ether<br>$C_{12}H_5Cl_5O$<br>(PCDE-100)<br>[104294-16-8]<br>FONWDRSQXQZNBN-UHFFFAOYSA-N | $2.1\times10^{-2}$<br>$1.3\times10^{-2}$ | <br>5800 | Kurz and Ballschmiter (1999)<br>Paasivirta et al. (1999) | V<br>T | |
| 2,2',4,5,5'-pentachlorodiphenyl ether<br>$C_{12}H_5Cl_5O$<br>(PCDE-101)<br>[131138-21-1]<br>MKIAKZAWONVRFF-UHFFFAOYSA-N | $1.6\times10^{-2}$ | | Kurz and Ballschmiter (1999) | V | |
| 2,2',4,5,6'-pentachlorodiphenyl ether<br>$C_{12}H_5Cl_5O$<br>(PCDE-102)<br>[130892-67-0]<br>BOJBTUWEQMUWHT-UHFFFAOYSA-N | $3.7\times10^{-2}$ | | Kurz and Ballschmiter (1999) | V | |



Table A6.5: Polychlorinated diphenyl ethers (PCDEs) (... continued)

| Substance<br>Formula<br>(Trivial Name)<br>[CAS Registry Number]<br>InChIKey | $H_s^{cp}$<br>(at $T^\ominus$)<br><br>$\left[\dfrac{\text{mol}}{\text{m}^3\,\text{Pa}}\right]$ | $\dfrac{\text{d}\ln H_s^{cp}}{\text{d}(1/T)}$<br><br>[K] | Reference | Type | Note |
|---|---|---|---|---|---|
| 2,3,3',4,4'-pentachlorodiphenyl ether<br>$C_{12}H_5Cl_5O$<br>(PCDE-105)<br>[85918-31-6]<br>PKEGIXYNAGGYPN-UHFFFAOYSA-N | $4.2\times10^{-2}$ | | Kurz and Ballschmiter (1999) | V | |
| 2,3,3',4,5'-pentachlorodiphenyl ether<br>$C_{12}H_5Cl_5O$<br>(PCDE-108)<br>[160282-07-5]<br>HCTKORQPUZCIAO-UHFFFAOYSA-N | $1.5\times10^{-2}$ | | Kurz and Ballschmiter (1999) | V | |
| 2,3,3',4,6-pentachlorodiphenyl ether<br>$C_{12}H_5Cl_5O$<br>(PCDE-109)<br>[727738-64-9]<br>SEIXDCGLHKOBEO-UHFFFAOYSA-N | $1.4\times10^{-2}$ | | Kurz and Ballschmiter (1999) | V | |
| 2,3,3',4',6-pentachlorodiphenyl ether<br>$C_{12}H_5Cl_5O$<br>(PCDE-110)<br>[159553-69-2]<br>KASGFDPQAMDPAZ-UHFFFAOYSA-N | $3.6\times10^{-2}$ | | Kurz and Ballschmiter (1999) | V | |
| 2,3,4,4',5-pentachlorodiphenyl ether<br>$C_{12}H_5Cl_5O$<br>(PCDE-114)<br>[113464-17-8]<br>WSTVMACYLXUQIM-UHFFFAOYSA-N | $1.2\times10^{-2}$ | | Kurz and Ballschmiter (1999) | V | |
| 2,3,4,4',6-pentachlorodiphenyl ether<br>$C_{12}H_5Cl_5O$<br>(PCDE-115)<br>[160282-05-3]<br>LUNNFBBTEHRYMW-UHFFFAOYSA-N | $1.1\times10^{-2}$ | | Kurz and Ballschmiter (1999) | V | |
| 2,3,4,5,6-pentachlorodiphenyl ether<br>$C_{12}H_5Cl_5O$<br>(PCDE-116)<br>[22274-42-6]<br>JDWOFUWJURZFFF-UHFFFAOYSA-N | $6.6\times10^{-3}$ | | Kurz and Ballschmiter (1999) | V | |



Table A6.5: Polychlorinated diphenyl ethers (PCDEs) (...continued)

| Substance Formula (Trivial Name) [CAS Registry Number] InChIKey | $H_s^{cp}$ (at $T^{\ominus}$) $\left[\dfrac{\mathrm{mol}}{\mathrm{m}^3\,\mathrm{Pa}}\right]$ | $\dfrac{\mathrm{d}\ln H_s^{cp}}{\mathrm{d}(1/T)}$ [K] | Reference | Type | Note |
|---|---|---|---|---|---|
| 2,3,4',5,6-pentachlorodiphenyl ether $C_{12}H_5Cl_5O$ (PCDE-117) [63646-55-9] UETIKWAWMZLQQZ-UHFFFAOYSA-N | $1.1\times10^{-2}$ | | Kurz and Ballschmiter (1999) | V | |
| 2,3',4,4',5-pentachlorodiphenyl ether $C_{12}H_5Cl_5O$ (PCDE-118) [60123-65-1] NRAVUGAGYOBXIG-UHFFFAOYSA-N | $1.5\times10^{-2}$ | | Kurz and Ballschmiter (1999) | V | |
| 2,3',4,4',6-pentachlorodiphenyl ether $C_{12}H_5Cl_5O$ (PCDE-119) [157683-74-4] POOXVKSDWCXIBU-UHFFFAOYSA-N | $1.3\times10^{-2}$ | | Kurz and Ballschmiter (1999) | V | |
| 2,3',4,5,5'-pentachlorodiphenyl ether $C_{12}H_5Cl_5O$ (PCDE-120) [160282-04-2] XIFRTJZLSYOLIU-UHFFFAOYSA-N | $4.8\times10^{-3}$ | | Kurz and Ballschmiter (1999) | V | |
| 2,3',4,4',5'-pentachlorodiphenyl ether $C_{12}H_5Cl_5O$ (PCDE-123) [160282-06-4] BUAXMSGDEZZOJD-UHFFFAOYSA-N | $1.3\times10^{-2}$ | | Kurz and Ballschmiter (1999) | V | |
| 3,3',4,4',5-pentachlorodiphenyl ether $C_{12}H_5Cl_5O$ (PCDE-126) [94339-59-0] WDBLKMRZRGLISL-UHFFFAOYSA-N | $1.0\times10^{-2}$ | | Kurz and Ballschmiter (1999) | V | |
| 2,2',3,3',4,4'-hexachlorodiphenyl ether $C_{12}H_4Cl_6O$ (PCDE-128) [71585-39-2] GSWMXIIUDJOXNF-UHFFFAOYSA-N | $8.3\times10^{-2}$ | | Kurz and Ballschmiter (1999) | V | |



Table A6.5: Polychlorinated diphenyl ethers (PCDEs) (...continued)

| Substance Formula (Trivial Name) [CAS Registry Number] InChIKey | $H_s^{cp}$ (at $T^\ominus$) $\left[\dfrac{\text{mol}}{\text{m}^3\,\text{Pa}}\right]$ | $\dfrac{\text{d}\ln H_s^{cp}}{\text{d}(1/T)}$ [K] | Reference | Type | Note |
|---|---|---|---|---|---|
| 2,2',3,3',4,5'-hexachlorodiphenyl ether C$_{12}$H$_4$Cl$_6$O (PCDE-130) [76621-14-2] JTEWACHFMNTRBB-UHFFFAOYSA-N | $1.5\times10^{-2}$ | | Kurz and Ballschmiter (1999) | V | |
| 2,2',3,3',4,6'-hexachlorodiphenyl ether C$_{12}$H$_4$Cl$_6$O (PCDE-132) [124076-66-0] YDVYLSIVXYLBBX-UHFFFAOYSA-N | $6.2\times10^{-2}$ | | Kurz and Ballschmiter (1999) | V | |
| 2,2',3,4,4',5-hexachlorodiphenyl ether C$_{12}$H$_4$Cl$_6$O (PCDE-137) [71585-38-1] LDPJEMNNPDFBCQ-UHFFFAOYSA-N | $1.8\times10^{-2}$ $1.9\times10^{-2}$ | 6400 | Kurz and Ballschmiter (1999) Paasivirta et al. (1999) | V T | |
| 2,2',3,4,4',5'-hexachlorodiphenyl ether C$_{12}$H$_4$Cl$_6$O (PCDE-138) [71585-36-9] PHSJJYZIFPWCLZ-UHFFFAOYSA-N | $2.9\times10^{-2}$ $2.8\times10^{-2}$ | 6500 | Kurz and Ballschmiter (1999) Paasivirta et al. (1999) | V T | |
| 2,2',3,4,4',6-hexachlorodiphenyl ether C$_{12}$H$_4$Cl$_6$O (PCDE-139) [106220-83-1] BIWCEXHDIQZFHI-UHFFFAOYSA-N | $9.8\times10^{-3}$ | | Kurz and Ballschmiter (1999) | V | |
| 2,2',3,4,4',6'-hexachlorodiphenyl ether C$_{12}$H$_4$Cl$_6$O (PCDE-140) [106220-82-0] XDAOZRZWYVKIAK-UHFFFAOYSA-N | $3.0\times10^{-2}$ | | Kurz and Ballschmiter (1999) | V | |
| 2,2',3,4',5,5'-hexachlorodiphenyl ether C$_{12}$H$_4$Cl$_6$O (PCDE-146) [162853-28-3] SFTBUGPDOJDENF-UHFFFAOYSA-N | $1.0\times10^{-2}$ | | Kurz and Ballschmiter (1999) | V | |





Table A6.5: Polychlorinated diphenyl ethers (PCDEs) (. . . continued)

| Substance<br>Formula<br>(Trivial Name)<br>[CAS Registry Number]<br>InChIKey | $H_s^{cp}$<br>(at $T^{\ominus}$)<br>$\left[\dfrac{\mathrm{mol}}{\mathrm{m^3\,Pa}}\right]$ | $\dfrac{\mathrm{d}\ln H_s^{cp}}{\mathrm{d}(1/T)}$<br><br>[K] | Reference | Type | Note |
|---|---|---|---|---|---|
| 2,2',3,4',5,6-hexachlorodiphenyl ether<br>$C_{12}H_4Cl_6O$<br>(PCDE-147)<br>[116995-18-7]<br>DIBYIHMYEPMGKC-UHFFFAOYSA-N | $1.0\times10^{-2}$ | | Kurz and Ballschmiter (1999) | V | |
| 2,2',3,4',5',6-hexachlorodiphenyl ether<br>$C_{12}H_4Cl_6O$<br>(PCDE-149)<br>[85918-37-2]<br>RDKSFIZNZGWNLA-UHFFFAOYSA-N | $3.1\times10^{-2}$ | | Kurz and Ballschmiter (1999) | V | |
| 2,2',4,4',5,5'-hexachlorodiphenyl ether<br>$C_{12}H_4Cl_6O$<br>(PCDE-153)<br>[71859-30-8]<br>PECXRRMHOQBOIE-UHFFFAOYSA-N | $1.3\times10^{-2}$<br>$1.1\times10^{-2}$ | <br>6300 | Kurz and Ballschmiter (1999)<br>Paasivirta et al. (1999) | V<br>T | |
| 2,2',4,4',5,6'-hexachlorodiphenyl ether<br>$C_{12}H_4Cl_6O$<br>(PCDE-154)<br>[106220-81-9]<br>XKQUAQGQLZDDBV-UHFFFAOYSA-N | $1.4\times10^{-2}$<br>$4.4\times10^{-3}$ | <br>5900 | Kurz and Ballschmiter (1999)<br>Paasivirta et al. (1999) | V<br>T | |
| 2,3,3',4,4',5-hexachlorodiphenyl ether<br>$C_{12}H_4Cl_6O$<br>(PCDE-156)<br>[109828-22-0]<br>GHTWMHLFFDEYRE-UHFFFAOYSA-N | $1.2\times10^{-2}$ | | Kurz and Ballschmiter (1999) | V | |
| 2,3,3',4,4',5'-hexachlorodiphenyl ether<br>$C_{12}H_4Cl_6O$<br>(PCDE-157)<br>[94339-60-3]<br>FEIDIWDEJQVIPO-UHFFFAOYSA-N | $2.8\times10^{-2}$ | | Kurz and Ballschmiter (1999) | V | |
| 2,3,3',4',5,6-hexachlorodiphenyl ether<br>$C_{12}H_4Cl_6O$<br>(PCDE-163)<br>[155999-97-6]<br>YLAYOOJLJKJGCF-UHFFFAOYSA-N | $1.6\times10^{-2}$ | | Kurz and Ballschmiter (1999) | V | |



Table A6.5: Polychlorinated diphenyl ethers (PCDEs) (. . . continued)

| Substance Formula (Trivial Name) [CAS Registry Number] InChIKey | $H_s^{cp}$ (at $T^\ominus$) $\left[\dfrac{\mathrm{mol}}{\mathrm{m}^3\,\mathrm{Pa}}\right]$ | $\dfrac{\mathrm{d}\ln H_s^{cp}}{\mathrm{d}(1/T)}$ [K] | Reference | Type | Note |
|---|---|---|---|---|---|
| 2,3,4,4',5,6-hexachlorodiphenyl ether $C_{12}H_4Cl_6O$ (PCDE-166) [63646-56-0] MXDRLBNLTLOWGV-UHFFFAOYSA-N | $5.0\times10^{-3}$ | | Kurz and Ballschmiter (1999) | V | |
| 2,3',4,4',5,5'-hexachlorodiphenyl ether $C_{12}H_4Cl_6O$ (PCDE-167) [131138-20-0] DYSUSQZPGJXVAL-UHFFFAOYSA-N | $8.3\times10^{-3}$ $9.0\times10^{-3}$ | 6200 | Kurz and Ballschmiter (1999) Paasivirta et al. (1999) | V T | |
| 2,2',3,3',4,4',5-heptachlorodiphenyl ether $C_{12}H_3Cl_7O$ (PCDE-170) [71585-40-5] BLBURLWSCHSPIS-UHFFFAOYSA-N | $2.0\times10^{-2}$ | | Kurz and Ballschmiter (1999) | V | |
| 2,2',3,3',4,5,6'-heptachlorodiphenyl ether $C_{12}H_3Cl_7O$ (PCDE-174) [159553-73-8] DQYYZAABYZAOOC-UHFFFAOYSA-N | $1.8\times10^{-2}$ | | Kurz and Ballschmiter (1999) | V | |
| 2,2',3,3',4,5',6'-heptachlorodiphenyl ether $C_{12}H_3Cl_7O$ (PCDE-177) [83992-71-6] RXPQVFMDOJTCKE-UHFFFAOYSA-N | $1.4\times10^{-2}$ | | Kurz and Ballschmiter (1999) | V | |
| 2,2',3,4,4',5,5'-heptachlorodiphenyl ether $C_{12}H_3Cl_7O$ (PCDE-180) [83992-69-2] YBYYCWMPXZRBNJ-UHFFFAOYSA-N | $5.0\times10^{-3}$ $1.9\times10^{-2}$ | 6800 | Kurz and Ballschmiter (1999) Paasivirta et al. (1999) | V T | |



Table A6.5: Polychlorinated diphenyl ethers (PCDEs) (. . . continued)

| Substance Formula (Trivial Name) [CAS Registry Number] InChIKey | $H_s^{cp}$ (at $T^{\ominus}$) $\left[\dfrac{\mathrm{mol}}{\mathrm{m^3\,Pa}}\right]$ | $\dfrac{\mathrm{d}\ln H_s^{cp}}{\mathrm{d}(1/T)}$ [K] | Reference | Type | Note |
|---|---|---|---|---|---|
| 2,2',3,4,4',5,6-heptachlorodiphenyl ether C$_{12}$H$_3$Cl$_7$O (PCDE-181) [157683-75-5] RPNFRXWGMTYOPA-UHFFFAOYSA-N | $3.4\times10^{-3}$ | | Kurz and Ballschmiter (1999) | V | |
| 2,2',3,4,4',5,6'-heptachlorodiphenyl ether C$_{12}$H$_3$Cl$_7$O (PCDE-182) [88467-63-4] XWRZWPPSQDHTFU-UHFFFAOYSA-N | $3.3\times10^{-3}$ | 6400 | Paasivirta et al. (1999) | T | |
| 2,2',3,4,4',6,6'-heptachlorodiphenyl ether C$_{12}$H$_3$Cl$_7$O (PCDE-184) [106220-84-2] WTYSJAGEYAAIFW-UHFFFAOYSA-N | $2.0\times10^{-1}$ | 7800 | Paasivirta et al. (1999) | T | |
| 2,2',3,4',5,5',6-heptachlorodiphenyl ether C$_{12}$H$_3$Cl$_7$O (PCDE-187) [109828-23-1] BHJCMMGUFRZLKK-UHFFFAOYSA-N | $7.4\times10^{-3}$ | | Kurz and Ballschmiter (1999) | V | |
| 2,3,3',4,4',5,5'-heptachlorodiphenyl ether C$_{12}$H$_3$Cl$_7$O (PCDE-189) [83992-72-7] JUHPMUOWCYBMEY-UHFFFAOYSA-N | $6.2\times10^{-3}$ | | Kurz and Ballschmiter (1999) | V | |
| 2,3,3',4,4',5,6-heptachlorodiphenyl ether C$_{12}$H$_3$Cl$_7$O (PCDE-190) [83992-70-5] QLSBRXLSQINWHM-UHFFFAOYSA-N | $5.8\times10^{-3}$ | | Kurz and Ballschmiter (1999) | V | |



Table A6.5: Polychlorinated diphenyl ethers (PCDEs) (. . . continued)

| Substance<br>Formula<br>(Trivial Name)<br>[CAS Registry Number]<br>InChIKey | $H_s^{cp}$<br>(at $T^\ominus$)<br>$\left[\dfrac{\text{mol}}{\text{m}^3\,\text{Pa}}\right]$ | $\dfrac{\mathrm{d}\ln H_s^{cp}}{\mathrm{d}(1/T)}$<br><br>[K] | Reference | Type | Note |
|---|---|---|---|---|---|
| 2,2',3,3',4,4',5,5'-<br>octachlorodiphenyl<br>ether<br>$C_{12}H_2Cl_8O$<br>(PCDE-194)<br>[57379-40-5]<br>IXZVOZCULZBCDY-UHFFFAOYSA-N | $4.3\times10^{-3}$ | | Kurz and Ballschmiter (1999) | V | |
| 2,2',3,3',4,4',5,6-<br>octachlorodiphenyl<br>ether<br>$C_{12}H_2Cl_8O$<br>(PCDE-195)<br>[65075-02-7]<br>YHUPMPHVHYBYJE-UHFFFAOYSA-N | $1.8\times10^{-3}$ | | Kurz and Ballschmiter (1999) | V | |
| 2,2',3,3',4,4',5,6'-<br>octachlorodiphenyl<br>ether<br>$C_{12}H_2Cl_8O$<br>(PCDE-196)<br>[85918-38-3]<br>HYNYIQBCBHJTOZ-UHFFFAOYSA-N | $8.7\times10^{-3}$ | 7100 | Paasivirta et al. (1999) | T | |
| 2,2',3,3',4,4',6,6'-<br>octachlorodiphenyl<br>ether<br>$C_{12}H_2Cl_8O$<br>(PCDE-197)<br>[117948-62-6]<br>GRWRDFGBFYSOKR-UHFFFAOYSA-N | $7.7\times10^{-3}$ | 7000 | Paasivirta et al. (1999) | T | |
| 2,2',3,3',4,5,5',6'-<br>octachlorodiphenyl<br>ether<br>$C_{12}H_2Cl_8O$<br>(PCDE-199)<br>[83992-76-1]<br>MRHQOXCYYPFGEE-UHFFFAOYSA-N | $2.6\times10^{-3}$ | | Kurz and Ballschmiter (1999) | V | |
| 2,2',3,4,4',5,5',6-<br>octachlorodiphenyl<br>ether<br>$C_{12}H_2Cl_8O$<br>(PCDE-203)<br>[83992-75-0]<br>PKOSPVZTRLMBSK-UHFFFAOYSA-N | $2.3\times10^{-3}$ | | Kurz and Ballschmiter (1999) | V | |





Table A6.5: Polychlorinated diphenyl ethers (PCDEs) (...continued)

| Substance<br>Formula<br>(Trivial Name)<br>[CAS Registry Number]<br><small>InChIKey</small> | $H_s^{cp}$<br>(at $T^\ominus$)<br>$\left[\dfrac{\text{mol}}{\text{m}^3\,\text{Pa}}\right]$ | $\dfrac{\text{d}\ln H_s^{cp}}{\text{d}(1/T)}$<br><br>[K] | Reference | Type | Note |
|---|---|---|---|---|---|
| 2,2',3,3',4,4',5,5',6-<br>nonachlorodiphenyl<br>ether<br>$C_{12}HCl_9O$<br>(PCDE-206)<br>[83992-73-8]<br><small>FPEYJPVHPGDXDD-UHFFFAOYSA-N</small> | $5.1\times10^{-4}$ | | Kurz and Ballschmiter (1999) | V | |
| decachlorodiphenyl ether<br>$C_{12}Cl_{10}O$<br>(PCDE-209)<br>[31710-30-2]<br><small>CIPFDHFTBYJKQB-UHFFFAOYSA-N</small> | $7.1\times10^{-5}$ | | Kurz and Ballschmiter (1999) | V | |



### A6.6 Polychlorinated dibenzofuranes (PCDFs)

Table A6.6: Polychlorinated dibenzofuranes (PCDFs)

| Substance Formula (Trivial Name) [CAS Registry Number] InChIKey | $H_s^{cp}$ (at $T^{\ominus}$) $\left[\dfrac{\mathrm{mol}}{\mathrm{m^3\,Pa}}\right]$ | $\dfrac{\mathrm{d}\ln H_s^{cp}}{\mathrm{d}(1/T)}$ [K] | Reference | Type | Note |
|---|---|---|---|---|---|
| 1-chlorodibenzofuran $C_{12}H_7ClO$ (PCDF-1) [84761-86-4] WRSMJZYBNIAAEE-UHFFFAOYSA-N | $8.3\times10^{-2}$ | | Govers and Krop (1998) | Q | |
| 2-chlorodibenzofuran $C_{12}H_7ClO$ (PCDF-2) [51230-49-0] PRKTYWJFCODJOA-UHFFFAOYSA-N | $1.1\times10^{-1}$ | | Govers and Krop (1998) | Q | |
| 3-chlorodibenzofuran $C_{12}H_7ClO$ (PCDF-3) [25074-67-3] BBOZMMAURMEVAR-UHFFFAOYSA-N | $1.3\times10^{-1}$ | | Govers and Krop (1998) | Q | |
| 4-chlorodibenzofuran $C_{12}H_7ClO$ (PCDF-4) [74992-96-4] RHRYBWFAHXCUCR-UHFFFAOYSA-N | $8.9\times10^{-2}$ | | Govers and Krop (1998) | Q | |
| 1,2-dichlorodibenzofuran $C_{12}H_6Cl_2O$ (PCDF-12) [64126-85-8] QVAPRJWSOUITIF-UHFFFAOYSA-N | $1.5\times10^{-1}$ | | Govers and Krop (1998) | Q | |
| 1,3-dichlorodibenzofuran $C_{12}H_6Cl_2O$ (PCDF-13) [94538-00-8] VKIBKEFGJSPRJC-UHFFFAOYSA-N | $2.0\times10^{-1}$ | | Govers and Krop (1998) | Q | |
| 1,4-dichlorodibenzofuran $C_{12}H_6Cl_2O$ (PCDF-14) [94538-01-9] VHQCMZLPHWGUDB-UHFFFAOYSA-N | $1.5\times10^{-1}$ | | Govers and Krop (1998) | Q | |
| 1,6-dichlorodibenzofuran $C_{12}H_6Cl_2O$ (PCDF-16) [74992-97-5] JRSRWACZUFLWKF-UHFFFAOYSA-N | $1.4\times10^{-1}$ | | Govers and Krop (1998) | Q | |





Table A6.6: Polychlorinated dibenzofuranes (PCDFs) (... continued)

| Substance<br>Formula<br>(Trivial Name)<br>[CAS Registry Number]<br>InChIKey | $H_s^{cp}$<br>(at $T^\ominus$)<br>$\left[\dfrac{\mathrm{mol}}{\mathrm{m^3\,Pa}}\right]$ | $\dfrac{\mathrm{d}\ln H_s^{cp}}{\mathrm{d}(1/T)}$<br><br>[K] | Reference | Type | Note |
|---|---|---|---|---|---|
| 1,7-dichlorodibenzofuran<br>$C_{12}H_6Cl_2O$<br>(PCDF-17)<br>[94538-02-0]<br>XRQNHPGZRFKBJW-UHFFFAOYSA-N | $1.9\times10^{-1}$ | | Govers and Krop (1998) | Q | |
| 1,8-dichlorodibenzofuran<br>$C_{12}H_6Cl_2O$<br>(PCDF-18)<br>[81638-37-1]<br>UAFNQQBPUWFPGC-UHFFFAOYSA-N | $2.5\times10^{-1}$ | | Govers and Krop (1998) | Q | |
| 1,9-dichlorodibenzofuran<br>$C_{12}H_6Cl_2O$<br>(PCDF-19)<br>[70648-14-5]<br>AEKSGHBVWJHELW-UHFFFAOYSA-N | $2.0\times10^{-1}$ | | Govers and Krop (1998) | Q | |
| 2,3-dichlorodibenzofuran<br>$C_{12}H_6Cl_2O$<br>(PCDF-23)<br>[64126-86-9]<br>GETJJZZRPQFSFM-UHFFFAOYSA-N | $2.3\times10^{-1}$ | | Govers and Krop (1998) | Q | |
| 2,4-dichlorodibenzofuran<br>$C_{12}H_6Cl_2O$<br>(PCDF-24)<br>[24478-74-8]<br>LHTCKMYRNOGUOA-UHFFFAOYSA-N | $1.9\times10^{-1}$ | | Govers and Krop (1998) | Q | |
| 2,6-dichlorodibenzofuran<br>$C_{12}H_6Cl_2O$<br>(PCDF-26)<br>[60390-27-4]<br>XVLCNKFGFHUQNL-UHFFFAOYSA-N | $1.8\times10^{-1}$ | | Govers and Krop (1998) | Q | |
| 2,7-dichlorodibenzofuran<br>$C_{12}H_6Cl_2O$<br>(PCDF-27)<br>[74992-98-6]<br>DOZUTNBCPVUMPY-UHFFFAOYSA-N | $2.0\times10^{-1}$ | | Govers and Krop (1998) | Q | |
| 2,8-dichlorodibenzofuran<br>$C_{12}H_6Cl_2O$<br>(PCDF-28)<br>[5409-83-6]<br>IVVRJIDVYSPKFZ-UHFFFAOYSA-N | $1.6\times10^{-1}$<br>$1.6\times10^{-1}$<br>$1.6\times10^{-1}$<br>$4.2\times10^{-1}$<br>$2.6\times10^{-1}$<br>$2.2\times10^{-1}$ | | Duchowicz et al. (2020)<br>Mackay et al. (2006b)<br>Govers and Krop (1998)<br>Duchowicz et al. (2020)<br>Saçan et al. (2005)<br>Govers and Krop (1998) | V<br>V<br>V<br>Q<br>Q<br>Q | 186 |





Table A6.6: Polychlorinated dibenzofuranes (PCDFs) (...continued)

| Substance Formula (Trivial Name) [CAS Registry Number] InChIKey | $H_s^{cp}$ (at $T^\ominus$) $\left[\dfrac{\mathrm{mol}}{\mathrm{m}^3\,\mathrm{Pa}}\right]$ | $\dfrac{\mathrm{d}\ln H_s^{cp}}{\mathrm{d}(1/T)}$ [K] | Reference | Type | Note |
|---|---|---|---|---|---|
| 3,4-dichlorodibenzofuran $C_{12}H_6Cl_2O$ (PCDF-34) [94570-83-9] HYQGGNWDPQZLGX-UHFFFAOYSA-N | $1.9\times10^{-1}$ | | Govers and Krop (1998) | Q | |
| 3,6-dichlorodibenzofuran $C_{12}H_6Cl_2O$ (PCDF-36) [74918-40-4] DMOBGFAKTSXOHG-UHFFFAOYSA-N | $2.2\times10^{-1}$ | | Govers and Krop (1998) | Q | |
| 3,7-dichlorodibenzofuran $C_{12}H_6Cl_2O$ (PCDF-37) [58802-21-4] OLTYAGNRSNFHLK-UHFFFAOYSA-N | $3.0\times10^{-1}$ | | Govers and Krop (1998) | Q | |
| 4,6-dichlorodibenzofuran $C_{12}H_6Cl_2O$ (PCDF-46) [64560-13-0] BZFIZJNKNCJEAD-UHFFFAOYSA-N | $2.2\times10^{-1}$ | | Govers and Krop (1998) | Q | |
| 1,2,3-trichlorodibenzofuran $C_{12}H_5Cl_3O$ (PCDF-123) [83636-47-9] LADSWAAWPMGHBE-UHFFFAOYSA-N | $2.9\times10^{-1}$ | | Govers and Krop (1998) | Q | |
| 1,2,4-trichlorodibenzofuran $C_{12}H_5Cl_3O$ (PCDF-124) [24478-73-7] RQIWKWHVZLDXEJ-UHFFFAOYSA-N | $2.5\times10^{-1}$ | | Govers and Krop (1998) | Q | |
| 1,2,6-trichlorodibenzofuran $C_{12}H_5Cl_3O$ (PCDF-126) [64560-15-2] CYRZCBUWRUACKJ-UHFFFAOYSA-N | $2.3\times10^{-1}$ | | Govers and Krop (1998) | Q | |
| 1,2,7-trichlorodibenzofuran $C_{12}H_5Cl_3O$ (PCDF-127) [83704-37-4] AFOVQGPNOZDUEP-UHFFFAOYSA-N | $2.3\times10^{-1}$ | | Govers and Krop (1998) | Q | |





Table A6.6: Polychlorinated dibenzofuranes (PCDFs) (. . . continued)

| Substance Formula (Trivial Name) [CAS Registry Number] InChIKey | $H_s^{cp}$ (at $T^{\ominus}$) $\left[\dfrac{\text{mol}}{\text{m}^3\,\text{Pa}}\right]$ | $\dfrac{\text{d}\ln H_s^{cp}}{\text{d}(1/T)}$ [K] | Reference | Type | Note |
|---|---|---|---|---|---|
| 1,2,8-trichlorodibenzofuran $C_{12}H_5Cl_3O$ (PCDF-128) [83704-34-1] UYIGPSPCOXGBCS-UHFFFAOYSA-N | $3.9\times10^{-1}$ | | Govers and Krop (1998) | Q | |
| 1,2,9-trichlorodibenzofuran $C_{12}H_5Cl_3O$ (PCDF-129) [83704-38-5] ICTXINQOXPMQPM-UHFFFAOYSA-N | $4.8\times10^{-1}$ | | Govers and Krop (1998) | Q | |
| 1,3,4-trichlorodibenzofuran $C_{12}H_5Cl_3O$ (PCDF-134) [82911-61-3] FIPOITUDJFQRRN-UHFFFAOYSA-N | $2.8\times10^{-1}$ | | Govers and Krop (1998) | Q | |
| 1,3,6-trichlorodibenzofuran $C_{12}H_5Cl_3O$ (PCDF-136) [83704-39-6] AGPTXRLVBIRYLV-UHFFFAOYSA-N | $3.3\times10^{-1}$ | | Govers and Krop (1998) | Q | |
| 1,3,7-trichlorodibenzofuran $C_{12}H_5Cl_3O$ (PCDF-137) [64560-16-3] INRBXOGZPDFCLQ-UHFFFAOYSA-N | $4.1\times10^{-1}$ | | Govers and Krop (1998) | Q | |
| 1,3,8-trichlorodibenzofuran $C_{12}H_5Cl_3O$ (PCDF-138) [76621-12-0] PHFSTDOPTZHECA-UHFFFAOYSA-N | $4.2\times10^{-1}$ | | Govers and Krop (1998) | Q | |
| 1,3,9-trichlorodibenzofuran $C_{12}H_5Cl_3O$ (PCDF-139) [83704-40-9] SPPGDLROKBFNNO-UHFFFAOYSA-N | $4.4\times10^{-1}$ | | Govers and Krop (1998) | Q | |
| 1,4,6-trichlorodibenzofuran $C_{12}H_5Cl_3O$ (PCDF-146) [82911-60-2] XDQRWSUJOURTHK-UHFFFAOYSA-N | $3.5\times10^{-1}$ | | Govers and Krop (1998) | Q | |



Table A6.6: Polychlorinated dibenzofuranes (PCDFs) (...continued)

| Substance Formula (Trivial Name) [CAS Registry Number] InChIKey | $H_s^{cp}$ (at $T^{\ominus}$) $\left[\dfrac{\text{mol}}{\text{m}^3\,\text{Pa}}\right]$ | $\dfrac{\text{d}\ln H_s^{cp}}{\text{d}(1/T)}$ [K] | Reference | Type | Note |
|---|---|---|---|---|---|
| 1,4,7-trichlorodibenzofuran $C_{12}H_5Cl_3O$ (PCDF-147) [83704-41-0] XTLMVJUSIANHND-UHFFFAOYSA-N | $3.2\times10^{-1}$ | | Govers and Krop (1998) | Q | |
| 1,4,8-trichlorodibenzofuran $C_{12}H_5Cl_3O$ (PCDF-148) [64560-14-1] PCQCYDNFBUADOK-UHFFFAOYSA-N | $3.9\times10^{-1}$ | | Govers and Krop (1998) | Q | |
| 1,4,9-trichlorodibenzofuran $C_{12}H_5Cl_3O$ (PCDF-149) [70648-13-4] NVYUPBVXKLTYHV-UHFFFAOYSA-N | $3.5\times10^{-1}$ | | Govers and Krop (1998) | Q | |
| 1,6,7-trichlorodibenzofuran $C_{12}H_5Cl_3O$ (PCDF-167) [83704-46-5] HEWLCNCSWBMZHG-UHFFFAOYSA-N | $2.7\times10^{-1}$ | | Govers and Krop (1998) | Q | |
| 1,6,8-trichlorodibenzofuran $C_{12}H_5Cl_3O$ (PCDF-168) [82911-59-9] ZOBVYDQWZXUJNO-UHFFFAOYSA-N | $4.1\times10^{-1}$ | | Govers and Krop (1998) | Q | |
| 1,7,8-trichlorodibenzofuran $C_{12}H_5Cl_3O$ (PCDF-178) [58802-18-9] YNMCXGLWHTVMQO-UHFFFAOYSA-N | $4.6\times10^{-1}$ | | Govers and Krop (1998) | Q | |
| 2,3,4-trichlorodibenzofuran $C_{12}H_5Cl_3O$ (PCDF-234) [57117-34-7] YDLADWPBKZODCJ-UHFFFAOYSA-N | $3.1\times10^{-1}$ | | Govers and Krop (1998) | Q | |
| 2,3,6-trichlorodibenzofuran $C_{12}H_5Cl_3O$ (PCDF-236) [57117-33-6] CPMGJTLNRBIKQM-UHFFFAOYSA-N | $3.4\times10^{-1}$ | | Govers and Krop (1998) | Q | |



Table A6.6: Polychlorinated dibenzofuranes (PCDFs) (. . . continued)

| Substance<br>Formula<br>(Trivial Name)<br>[CAS Registry Number]<br>InChIKey | $H_s^{cp}$<br>(at $T^\ominus$)<br>$\left[ \dfrac{\text{mol}}{\text{m}^3\,\text{Pa}} \right]$ | $\dfrac{\text{d}\ln H_s^{cp}}{\text{d}(1/T)}$<br><br>[K] | Reference | Type | Note |
|---|---|---|---|---|---|
| 2,3,7-trichlorodibenzofuran<br>$C_{12}H_5Cl_3O$<br>(PCDF-237)<br>[58802-17-8]<br>CKXMNTLGGAOERF-UHFFFAOYSA-N | $3.5\times10^{-1}$ | | Govers and Krop (1998) | Q | |
| 2,3,8-trichlorodibenzofuran<br>$C_{12}H_5Cl_3O$<br>(PCDF-238)<br>[57117-32-5]<br>NUNSNNOYACKRIK-UHFFFAOYSA-N | $3.1\times10^{-1}$ | | Govers and Krop (1998) | Q | |
| 2,4,6-trichlorodibenzofuran<br>$C_{12}H_5Cl_3O$<br>(PCDF-246)<br>[58802-14-5]<br>GTKURBTWZFQHHY-UHFFFAOYSA-N | $4.2\times10^{-1}$ | | Govers and Krop (1998) | Q | |
| 2,4,7-trichlorodibenzofuran<br>$C_{12}H_5Cl_3O$<br>(PCDF-247)<br>[83704-42-1]<br>SZBZRVJHPZIGAY-UHFFFAOYSA-N | $3.1\times10^{-1}$ | | Govers and Krop (1998) | Q | |
| 2,4,8-trichlorodibenzofuran<br>$C_{12}H_5Cl_3O$<br>(PCDF-248)<br>[54589-71-8]<br>WJURXKWTCOMRCE-UHFFFAOYSA-N | $3.2\times10^{-1}$ | | Govers and Krop (1998) | Q | |
| 3,4,6-trichlorodibenzofuran<br>$C_{12}H_5Cl_3O$<br>(PCDF-346)<br>[83704-43-2]<br>XEUHBCMRIXEBTD-UHFFFAOYSA-N | $4.3\times10^{-1}$ | | Govers and Krop (1998) | Q | |
| 3,4,7-trichlorodibenzofuran<br>$C_{12}H_5Cl_3O$<br>(PCDF-347)<br>[83704-44-3]<br>RPJJBTSXPNFUTA-UHFFFAOYSA-N | $3.9\times10^{-1}$ | | Govers and Krop (1998) | Q | |
| 3,4,8-trichlorodibenzofuran<br>$C_{12}H_5Cl_3O$<br>(PCDF-348)<br>[83704-45-4]<br>YTZWNIQGGIJHJX-UHFFFAOYSA-N | $2.5\times10^{-1}$ | | Govers and Krop (1998) | Q | |





Table A6.6: Polychlorinated dibenzofuranes (PCDFs) (...continued)

| Substance<br>Formula<br>(Trivial Name)<br>[CAS Registry Number]<br>InChIKey | $H_s^{cp}$<br>(at $T^\ominus$)<br>$\left[\dfrac{\text{mol}}{\text{m}^3\,\text{Pa}}\right]$ | $\dfrac{\mathrm{d}\ln H_s^{cp}}{\mathrm{d}(1/T)}$<br><br>[K] | Reference | Type | Note |
|---|---|---|---|---|---|
| 1,2,3,4-tetrachlorodibenzofuran<br>$C_{12}H_4Cl_4O$<br>(PCDF-1234)<br>[24478-72-6]<br>AETAPIFVELRIDN-UHFFFAOYSA-N | $3.6\times10^{-1}$ | | Govers and Krop (1998) | Q | |
| 1,2,3,6-tetrachlorodibenzofuran<br>$C_{12}H_4Cl_4O$<br>(PCDF-1236)<br>[83704-21-6]<br>BBAXFLIBRPXRBP-UHFFFAOYSA-N | $4.1\times10^{-1}$ | | Govers and Krop (1998) | Q | |
| 1,2,3,7-tetrachlorodibenzofuran<br>$C_{12}H_4Cl_4O$<br>(PCDF-1237)<br>[83704-22-7]<br>MDNZFYGATDCKRB-UHFFFAOYSA-N | $3.9\times10^{-1}$ | | Govers and Krop (1998) | Q | |
| 1,2,3,8-tetrachlorodibenzofuran<br>$C_{12}H_4Cl_4O$<br>(PCDF-1238)<br>[62615-08-1]<br>KFQRHGKKSHJMCO-UHFFFAOYSA-N | $5.0\times10^{-1}$ | | Govers and Krop (1998) | Q | |
| 1,2,3,9-tetrachlorodibenzofuran<br>$C_{12}H_4Cl_4O$<br>(PCDF-1239)<br>[83704-23-8]<br>GNQHLJHTOATTOJ-UHFFFAOYSA-N | $7.9\times10^{-1}$ | | Govers and Krop (1998) | Q | |
| 1,2,4,6-tetrachlorodibenzofuran<br>$C_{12}H_4Cl_4O$<br>(PCDF-1246)<br>[71998-73-7]<br>NLVWEKPJFUPVMZ-UHFFFAOYSA-N | $5.1\times10^{-1}$ | | Govers and Krop (1998) | Q | |
| 1,2,4,7-tetrachlorodibenzofuran<br>$C_{12}H_4Cl_4O$<br>(PCDF-1247)<br>[83719-40-8]<br>MFURKMJLDQBHRA-UHFFFAOYSA-N | $3.5\times10^{-1}$ | | Govers and Krop (1998) | Q | |
| 1,2,4,8-tetrachlorodibenzofuran<br>$C_{12}H_4Cl_4O$<br>(PCDF-1248)<br>[64126-87-0]<br>BFTASCFRRBHFFK-UHFFFAOYSA-N | $5.5\times10^{-1}$ | | Govers and Krop (1998) | Q | |



Table A6.6: Polychlorinated dibenzofuranes (PCDFs) (...continued)

| Substance<br>Formula<br>(Trivial Name)<br>[CAS Registry Number]<br>InChIKey | $H_s^{cp}$<br>(at $T^\ominus$)<br>$\left[\dfrac{\mathrm{mol}}{\mathrm{m}^3\,\mathrm{Pa}}\right]$ | $\dfrac{\mathrm{d}\ln H_s^{cp}}{\mathrm{d}(1/T)}$<br><br>[K] | Reference | Type | Note |
|---|---|---|---|---|---|
| 1,2,4,9-tetrachlorodibenzofuran<br>$C_{12}H_4Cl_4O$<br>(PCDF-1249)<br>[83704-24-9]<br>ZUYYTCCHKQGGOV-UHFFFAOYSA-N | $7.4\times10^{-1}$ | | Govers and Krop (1998) | Q | |
| 1,2,6,7-tetrachlorodibenzofuran<br>$C_{12}H_4Cl_4O$<br>(PCDF-1267)<br>[83704-25-0]<br>AUEWYHHKDYUYMI-UHFFFAOYSA-N | $2.8\times10^{-1}$ | | Govers and Krop (1998) | Q | |
| 1,2,6,8-tetrachlorodibenzofuran<br>$C_{12}H_4Cl_4O$<br>(PCDF-1268)<br>[83710-07-0]<br>KCVGVSIBBGIJNZ-UHFFFAOYSA-N | $5.5\times10^{-1}$ | | Govers and Krop (1998) | Q | |
| 1,2,6,9-tetrachlorodibenzofuran<br>$C_{12}H_4Cl_4O$<br>(PCDF-1269)<br>[70648-18-9]<br>IEPGLLVEBKXASE-UHFFFAOYSA-N | $7.1\times10^{-1}$ | | Govers and Krop (1998) | Q | |
| 1,2,7,8-tetrachlorodibenzofuran<br>$C_{12}H_4Cl_4O$<br>(PCDF-1278)<br>[58802-20-3]<br>JODWPAQNABOHDG-UHFFFAOYSA-N | $1.1$<br>$4.8\times10^{-1}$ | | Saçan et al. (2005)<br>Govers and Krop (1998) | Q<br>Q | |
| 1,2,7,9-tetrachlorodibenzofuran<br>$C_{12}H_4Cl_4O$<br>(PCDF-1279)<br>[83704-26-1]<br>PDMFRPIFZAKMLH-UHFFFAOYSA-N | $6.9\times10^{-1}$ | | Govers and Krop (1998) | Q | |
| 1,2,8,9-tetrachlorodibenzofuran<br>$C_{12}H_4Cl_4O$<br>(PCDF-1289)<br>[70648-22-5]<br>OHYCQUKMNPHFPT-UHFFFAOYSA-N | $9.5\times10^{-1}$ | | Govers and Krop (1998) | Q | |
| 1,3,4,6-tetrachlorodibenzofuran<br>$C_{12}H_4Cl_4O$<br>(PCDF-1346)<br>[83704-27-2]<br>CBBDONKZEJKFJP-UHFFFAOYSA-N | $6.3\times10^{-1}$ | | Govers and Krop (1998) | Q | |



Table A6.6: Polychlorinated dibenzofuranes (PCDFs) (. . . continued)

| Substance Formula (Trivial Name) [CAS Registry Number] InChIKey | $H_s^{cp}$ (at $T^\ominus$) $\left[\dfrac{\text{mol}}{\text{m}^3\,\text{Pa}}\right]$ | $\dfrac{\text{d}\ln H_s^{cp}}{\text{d}(1/T)}$ [K] | Reference | Type | Note |
|---|---|---|---|---|---|
| 1,3,4,7-tetrachlorodibenzofuran $C_{12}H_4Cl_4O$ (PCDF-1347) [70648-16-7] UMKCVDZTSZNWRH-UHFFFAOYSA-N | $5.1\times10^{-1}$ | | Govers and Krop (1998) | Q | |
| 1,3,4,8-tetrachlorodibenzofuran $C_{12}H_4Cl_4O$ (PCDF-1348) [92341-04-3] XWKRJSMNDOXIHS-UHFFFAOYSA-N | $5.0\times10^{-1}$ | | Govers and Krop (1998) | Q | |
| 1,3,4,9-tetrachlorodibenzofuran $C_{12}H_4Cl_4O$ (PCDF-1349) [83704-28-3] PNCYJHOTZSKZPA-UHFFFAOYSA-N | $5.8\times10^{-1}$ | | Govers and Krop (1998) | Q | |
| 1,3,6,7-tetrachlorodibenzofuran $C_{12}H_4Cl_4O$ (PCDF-1367) [57117-36-9] PITDPGCTIMCYEZ-UHFFFAOYSA-N | $5.1\times10^{-1}$ | | Govers and Krop (1998) | Q | |
| 1,3,6,8-tetrachlorodibenzofuran $C_{12}H_4Cl_4O$ (PCDF-1368) [71998-72-6] BDXKVABZWRZKOS-UHFFFAOYSA-N | $6.2\times10^{-1}$ | | Govers and Krop (1998) | Q | |
| 1,3,6,9-tetrachlorodibenzofuran $C_{12}H_4Cl_4O$ (PCDF-1369) [83690-98-6] NJQQZRLWVLWNGD-UHFFFAOYSA-N | $6.8\times10^{-1}$ | | Govers and Krop (1998) | Q | |
| 1,3,7,8-tetrachlorodibenzofuran $C_{12}H_4Cl_4O$ (PCDF-1378) [57117-35-8] CSXDVUGDFSYXTD-UHFFFAOYSA-N | $6.5\times10^{-1}$ | | Govers and Krop (1998) | Q | |
| 1,3,7,9-tetrachlorodibenzofuran $C_{12}H_4Cl_4O$ (PCDF-1379) [64560-17-4] IEMJMCVYAORWJV-UHFFFAOYSA-N | $7.9\times10^{-1}$ | | Govers and Krop (1998) | Q | |





Table A6.6: Polychlorinated dibenzofuranes (PCDFs) (...continued)

| Substance<br>Formula<br>(Trivial Name)<br>[CAS Registry Number]<br>InChIKey | $H_s^{cp}$<br>(at $T^{\ominus}$)<br><br>$\left[\dfrac{\text{mol}}{\text{m}^3\,\text{Pa}}\right]$ | $\dfrac{\mathrm{d}\ln H_s^{cp}}{\mathrm{d}(1/T)}$<br><br>[K] | Reference | Type | Note |
|---|---|---|---|---|---|
| 1,4,6,7-tetrachlorodibenzofuran<br>$C_{12}H_4Cl_4O$<br>(PCDF-1467)<br>[66794-59-0]<br>RBFNYMHNIOKXLA-UHFFFAOYSA-N | $5.9\times10^{-1}$ | | Govers and Krop (1998) | Q | |
| 1,4,6,8-tetrachlorodibenzofuran<br>$C_{12}H_4Cl_4O$<br>(PCDF-1468)<br>[82911-58-8]<br>VHOBVPNETFRJED-UHFFFAOYSA-N | $8.5\times10^{-1}$ | | Govers and Krop (1998) | Q | |
| 1,4,6,9-tetrachlorodibenzofuran<br>$C_{12}H_4Cl_4O$<br>(PCDF-1469)<br>[70648-19-0]<br>JAYSBJFHDJOMGZ-UHFFFAOYSA-N | $7.9\times10^{-1}$ | | Govers and Krop (1998) | Q | |
| 1,4,7,8-tetrachlorodibenzofuran<br>$C_{12}H_4Cl_4O$<br>(PCDF-1478)<br>[83704-29-4]<br>IVRDRRWQABCXLY-UHFFFAOYSA-N | $6.5\times10^{-1}$ | | Govers and Krop (1998) | Q | |
| 1,6,7,8-tetrachlorodibenzofuran<br>$C_{12}H_4Cl_4O$<br>(PCDF-1678)<br>[83704-33-0]<br>KOJMOXYETDLOPN-UHFFFAOYSA-N | $5.8\times10^{-1}$ | | Govers and Krop (1998) | Q | |
| 2,3,4,6-tetrachlorodibenzofuran<br>$C_{12}H_4Cl_4O$<br>(PCDF-2346)<br>[83704-30-7]<br>JNVHSHPAAMRSKK-UHFFFAOYSA-N | $6.2\times10^{-1}$ | | Govers and Krop (1998) | Q | |
| 2,3,4,7-tetrachlorodibenzofuran<br>$C_{12}H_4Cl_4O$<br>(PCDF-2347)<br>[83704-31-8]<br>BROFYOSLFHTGCQ-UHFFFAOYSA-N | $4.4\times10^{-1}$ | | Govers and Krop (1998) | Q | |
| 2,3,4,8-tetrachlorodibenzofuran<br>$C_{12}H_4Cl_4O$<br>(PCDF-2348)<br>[83704-32-9]<br>IIKXRERKNIJJQY-UHFFFAOYSA-N | $3.5\times10^{-1}$ | | Govers and Krop (1998) | Q | |



Table A6.6: Polychlorinated dibenzofuranes (PCDFs) (...continued)

| Substance<br>Formula<br>(Trivial Name)<br>[CAS Registry Number]<br>InChIKey | $H_s^{cp}$<br>(at $T^{\ominus}$)<br>$\left[\dfrac{\text{mol}}{\text{m}^3\,\text{Pa}}\right]$ | $\dfrac{\text{d}\ln H_s^{cp}}{\text{d}(1/T)}$<br><br>[K] | Reference | Type | Note |
|---|---|---|---|---|---|
| 2,3,6,7-tetrachlorodibenzofuran<br>$C_{12}H_4Cl_4O$<br>(PCDF-2367)<br>[57117-39-2]<br>MJCNKMDTLRSHNK-UHFFFAOYSA-N | $4.1\times10^{-1}$ | | Govers and Krop (1998) | Q | |
| 2,3,6,8-tetrachlorodibenzofuran<br>$C_{12}H_4Cl_4O$<br>(PCDF-2368)<br>[57117-37-0]<br>SPMZHWVRMWWTSY-UHFFFAOYSA-N | $4.2\times10^{-1}$ | | Govers and Krop (1998) | Q | |
| 2,3,7,8-tetrachlorodibenzofuran<br>$C_{12}H_4Cl_4O$<br>(PCDF-2378)<br>[51207-31-9]<br>KSMVNVHUTQZITP-UHFFFAOYSA-N | $5.9\times10^{-1}$<br>$6.8\times10^{-1}$<br>$8.5\times10^{-1}$<br>$2.2\times10^{-3}$<br>$2.4\times10^{-1}$<br>$1.2$<br>$6.4\times10^{-1}$<br>$7.2\times10^{-1}$<br>$3.7\times10^{-1}$<br>$5.9\times10^{-1}$ | <br><br><br>3700 | Friesen et al. (1993)<br>Mackay et al. (2006b)<br>Govers and Krop (1998)<br>Paasivirta et al. (1999)<br>Keshavarz et al. (2022)<br>Duchowicz et al. (2020)<br>HSDB (2015)<br>Saçan et al. (2005)<br>Govers and Krop (1998)<br>Duchowicz et al. (2020) | M<br>V<br>V<br>T<br>Q<br>Q<br>Q<br>Q<br>Q<br>? | <br><br><br><br><br><br>99<br><br><br>185, 21 |
| 2,4,6,7-tetrachlorodibenzofuran<br>$C_{12}H_4Cl_4O$<br>(PCDF-2467)<br>[57117-38-1]<br>NSDXKMRVQOSASS-UHFFFAOYSA-N | $5.4\times10^{-1}$ | | Govers and Krop (1998) | Q | |
| 2,4,6,8-tetrachlorodibenzofuran<br>$C_{12}H_4Cl_4O$<br>(PCDF-2468)<br>[58802-19-0]<br>ZSPJEACWUWSAGS-UHFFFAOYSA-N | $6.6\times10^{-1}$ | | Govers and Krop (1998) | Q | |
| 3,4,6,7-tetrachlorodibenzofuran<br>$C_{12}H_4Cl_4O$<br>(PCDF-3467)<br>[57117-40-5]<br>LMJLCLBSUNZLNW-UHFFFAOYSA-N | $7.1\times10^{-1}$ | | Govers and Krop (1998) | Q | |
| 1,2,3,4,6-pentachlorodibenzofuran<br>$C_{12}H_3Cl_5O$<br>(PCDF-12346)<br>[83704-47-6]<br>LIQJBAPSLUZUTB-UHFFFAOYSA-N | $7.1\times10^{-1}$ | | Govers and Krop (1998) | Q | |



Table A6.6: Polychlorinated dibenzofuranes (PCDFs) (. . . continued)

| Substance<br>Formula<br>(Trivial Name)<br>[CAS Registry Number]<br>InChIKey | $H_s^{cp}$<br>(at $T^\ominus$)<br>$\left[\dfrac{\text{mol}}{\text{m}^3\,\text{Pa}}\right]$ | $\dfrac{\mathrm{d}\ln H_s^{cp}}{\mathrm{d}(1/T)}$<br><br>[K] | Reference | Type | Note |
|---|---|---|---|---|---|
| 1,2,3,4,7-pentachlorodibenzofuran<br>$C_{12}H_3Cl_5O$<br>(PCDF-12347)<br>[83704-48-7]<br>DOJZTBGOWIYFAC-UHFFFAOYSA-N | $4.5\times10^{-1}$ | | Govers and Krop (1998) | Q | |
| 1,2,3,4,8-pentachlorodibenzofuran<br>$C_{12}H_3Cl_5O$<br>(PCDF-12348)<br>[67517-48-0]<br>ZCTNDJSCLPJCRA-UHFFFAOYSA-N | $5.5\times10^{-1}$ | | Govers and Krop (1998) | Q | |
| 1,2,3,4,9-pentachlorodibenzofuran<br>$C_{12}H_3Cl_5O$<br>(PCDF-12349)<br>[83704-49-8]<br>IQSZNZVZXOFRJS-UHFFFAOYSA-N | $9.8\times10^{-1}$ | | Govers and Krop (1998) | Q | |
| 1,2,3,6,7-pentachlorodibenzofuran<br>$C_{12}H_3Cl_5O$<br>(PCDF-12367)<br>[57117-42-7]<br>NZUPQBVDIWCPBX-UHFFFAOYSA-N | $4.2\times10^{-1}$ | | Govers and Krop (1998) | Q | |
| 1,2,3,6,8-pentachlorodibenzofuran<br>$C_{12}H_3Cl_5O$<br>(PCDF-12368)<br>[83704-51-2]<br>VHQJZOFUPXEZQZ-UHFFFAOYSA-N | $6.5\times10^{-1}$ | | Govers and Krop (1998) | Q | |
| 1,2,3,6,9-pentachlorodibenzofuran<br>$C_{12}H_3Cl_5O$<br>(PCDF-12369)<br>[83704-52-3]<br>IPSFQAKXDJVQOX-UHFFFAOYSA-N | $1.1$ | | Govers and Krop (1998) | Q | |
| 1,2,3,7,8-pentachlorodibenzofuran<br>$C_{12}H_3Cl_5O$<br>(PCDF-12378)<br>[57117-41-6]<br>SBMIVUVRFPGOEB-UHFFFAOYSA-N | $8.7\times10^{-4}$<br>$5.2\times10^{-1}$ | 3000 | Paasivirta et al. (1999)<br>Govers and Krop (1998) | T<br>Q | |
| 1,2,3,7,9-pentachlorodibenzofuran<br>$C_{12}H_3Cl_5O$<br>(PCDF-12379)<br>[83704-53-4]<br>JVUSEQPOWCBYNG-UHFFFAOYSA-N | $1.0$ | | Govers and Krop (1998) | Q | |



Table A6.6: Polychlorinated dibenzofuranes (PCDFs) (. . . continued)

| Substance<br>Formula<br>(Trivial Name)<br>[CAS Registry Number]<br>InChIKey | $H_s^{cp}$<br>(at $T^\ominus$)<br>$\left[\dfrac{\text{mol}}{\text{m}^3\,\text{Pa}}\right]$ | $\dfrac{\text{d}\ln H_s^{cp}}{\text{d}(1/T)}$<br><br>[K] | Reference | Type | Note |
|---|---|---|---|---|---|
| 1,2,3,8,9-pentachlorodibenzofuran<br>$C_{12}H_3Cl_5O$<br>(PCDF-12389)<br>[83704-54-5]<br>NKGLWUJPCTUDEH-UHFFFAOYSA-N | 2.0<br>1.1 | | Saçan et al. (2005)<br>Govers and Krop (1998) | Q<br>Q | |
| 1,2,4,6,7-pentachlorodibenzofuran<br>$C_{12}H_3Cl_5O$<br>(PCDF-12467)<br>[83704-50-1]<br>BQTCODOVGXJHRB-UHFFFAOYSA-N | $5.8\times10^{-1}$ | | Govers and Krop (1998) | Q | |
| 1,2,4,6,8-pentachlorodibenzofuran<br>$C_{12}H_3Cl_5O$<br>(PCDF-12468)<br>[69698-57-3]<br>JDTUAYPJSSXDNO-UHFFFAOYSA-N | 1.0 | | Govers and Krop (1998) | Q | |
| 1,2,4,6,9-pentachlorodibenzofuran<br>$C_{12}H_3Cl_5O$<br>(PCDF-12469)<br>[70648-24-7]<br>QDQFGVXGPSYEJU-UHFFFAOYSA-N | 1.5 | | Govers and Krop (1998) | Q | |
| 1,2,4,7,8-pentachlorodibenzofuran<br>$C_{12}H_3Cl_5O$<br>(PCDF-12478)<br>[58802-15-6]<br>GCFDWHIKRXCUPJ-UHFFFAOYSA-N | $6.2\times10^{-1}$ | | Govers and Krop (1998) | Q | |
| 1,2,4,7,9-pentachlorodibenzofuran<br>$C_{12}H_3Cl_5O$<br>(PCDF-12479)<br>[71998-74-8]<br>IYGVHNHWORPMFU-UHFFFAOYSA-N | $9.8\times10^{-1}$ | | Govers and Krop (1998) | Q | |
| 1,2,4,8,9-pentachlorodibenzofuran<br>$C_{12}H_3Cl_5O$<br>(PCDF-12489)<br>[70648-23-6]<br>ZSPAPWGNAGTCCA-UHFFFAOYSA-N | 1.3 | | Govers and Krop (1998) | Q | |
| 1,2,6,7,8-pentachlorodibenzofuran<br>$C_{12}H_3Cl_5O$<br>(PCDF-12678)<br>[69433-00-7]<br>ZAAVDGLJEOXCMV-UHFFFAOYSA-N | $5.2\times10^{-1}$ | | Govers and Krop (1998) | Q | |



Table A6.6: Polychlorinated dibenzofuranes (PCDFs) (...continued)

| Substance<br>Formula<br>(Trivial Name)<br>[CAS Registry Number]<br>InChIKey | $H_s^{cp}$<br>(at $T^{\ominus}$)<br><br>$\left[\dfrac{\text{mol}}{\text{m}^3\,\text{Pa}}\right]$ | $\dfrac{\text{d}\ln H_s^{cp}}{\text{d}(1/T)}$<br><br>[K] | Reference | Type | Note |
|---|---|---|---|---|---|
| 1,2,6,7,9-pentachlorodibenzofuran<br>$C_{12}H_3Cl_5O$<br>(PCDF-12679)<br>[70872-82-1]<br>QQRRQEVOSMEVQB-UHFFFAOYSA-N | $7.9 \times 10^{-1}$ | | Govers and Krop (1998) | Q | |
| 1,3,4,6,7-pentachlorodibenzofuran<br>$C_{12}H_3Cl_5O$<br>(PCDF-13467)<br>[83704-36-3]<br>JVYYDUWJIJMOGW-UHFFFAOYSA-N | $8.9 \times 10^{-1}$ | | Govers and Krop (1998) | Q | |
| 1,3,4,6,8-pentachlorodibenzofuran<br>$C_{12}H_3Cl_5O$<br>(PCDF-13468)<br>[83704-55-6]<br>COKIAYPRHYHYCH-UHFFFAOYSA-N | 1.0 | | Govers and Krop (1998) | Q | |
| 1,3,4,6,9-pentachlorodibenzofuran<br>$C_{12}H_3Cl_5O$<br>(PCDF-13469)<br>[70648-15-6]<br>MYSAFUYQSBEEDR-UHFFFAOYSA-N | 1.2 | | Govers and Krop (1998) | Q | |
| 1,3,4,7,8-pentachlorodibenzofuran<br>$C_{12}H_3Cl_5O$<br>(PCDF-13478)<br>[58802-16-7]<br>ORSUQGVCWLXKLZ-UHFFFAOYSA-N | $7.1 \times 10^{-1}$ | | Govers and Krop (1998) | Q | |
| 1,3,4,7,9-pentachlorodibenzofuran<br>$C_{12}H_3Cl_5O$<br>(PCDF-13479)<br>[70648-20-3]<br>GFXPLABVZOBCCW-UHFFFAOYSA-N | $9.5 \times 10^{-1}$ | | Govers and Krop (1998) | Q | |
| 1,3,6,7,8-pentachlorodibenzofuran<br>$C_{12}H_3Cl_5O$<br>(PCDF-13678)<br>[70648-21-4]<br>FRLMQDUYUJIHCZ-UHFFFAOYSA-N | $7.6 \times 10^{-1}$ | | Govers and Krop (1998) | Q | |
| 1,4,6,7,8-pentachlorodibenzofuran<br>$C_{12}H_3Cl_5O$<br>(PCDF-14678)<br>[83704-35-2]<br>VANGHZRYKXDPRR-UHFFFAOYSA-N | 1.1 | | Govers and Krop (1998) | Q | |





Table A6.6: Polychlorinated dibenzofuranes (PCDFs) (. . . continued)

| Substance Formula (Trivial Name) [CAS Registry Number] InChIKey | $H_s^{cp}$ (at $T^\ominus$) $\left[\dfrac{\text{mol}}{\text{m}^3\,\text{Pa}}\right]$ | $\dfrac{\text{d}\ln H_s^{cp}}{\text{d}(1/T)}$ [K] | Reference | Type | Note |
|---|---|---|---|---|---|
| 2,3,4,6,7-pentachlorodibenzofuran $C_{12}H_3Cl_5O$ (PCDF-23467) [57117-43-8] SJFBZRQKGOGHEV-UHFFFAOYSA-N | $6.9\times10^{-1}$ | | Govers and Krop (1998) | Q | |
| 2,3,4,6,8-pentachlorodibenzofuran $C_{12}H_3Cl_5O$ (PCDF-23468) [67481-22-5] MKRFORPSRBMAIP-UHFFFAOYSA-N | $6.6\times10^{-1}$ | | Govers and Krop (1998) | Q | |
| 2,3,4,7,8-pentachlorodibenzofuran $C_{12}H_3Cl_5O$ (PCDF-23478) [57117-31-4] OGBQILNBLMPPDP-UHFFFAOYSA-N | 2.0<br>2.0<br>2.0<br>1.7<br>$2.3\times10^{-3}$<br>1.6<br>1.6<br>$3.9\times10^{-1}$ | <br><br><br><br>2900<br><br><br> | Duchowicz et al. (2020)<br>HSDB (2015)<br>Mackay et al. (2006b)<br>Govers and Krop (1998)<br>Paasivirta et al. (1999)<br>Duchowicz et al. (2020)<br>Saçan et al. (2005)<br>Govers and Krop (1998) | V<br>V<br>V<br>V<br>T<br>Q<br>Q<br>Q | 186 |
| 1,2,3,4,6,7-hexachlorodibenzofuran $C_{12}H_2Cl_6O$ (PCDF-123467) [79060-60-9] SNWFMKXFMVHBKD-UHFFFAOYSA-N | $6.8\times10^{-1}$ | | Govers and Krop (1998) | Q | |
| 1,2,3,4,6,8-hexachlorodibenzofuran $C_{12}H_2Cl_6O$ (PCDF-123468) [69698-60-8] UCFGNWHERVQWMZ-UHFFFAOYSA-N | $2.4\times10^{-4}$<br>$9.8\times10^{-1}$ | 2300<br> | Paasivirta et al. (1999)<br>Govers and Krop (1998) | T<br>Q | |
| 1,2,3,4,6,9-hexachlorodibenzofuran $C_{12}H_2Cl_6O$ (PCDF-123469) [91538-83-9] KFFUZIROJGXTQH-UHFFFAOYSA-N | 1.8 | | Govers and Krop (1998) | Q | |
| 1,2,3,4,7,8-hexachlorodibenzofuran $C_{12}H_2Cl_6O$ (PCDF-123478) [70648-26-9] LVYBAQIVPKCOEE-UHFFFAOYSA-N | $6.9\times10^{-1}$<br>$3.8\times10^{-1}$<br>$4.1\times10^{-4}$<br>2.0<br>$5.2\times10^{-1}$ | <br><br>2400<br><br> | Mackay et al. (2006b)<br>Govers and Krop (1998)<br>Paasivirta et al. (1999)<br>Saçan et al. (2005)<br>Govers and Krop (1998) | V<br>V<br>T<br>Q<br>Q | |





Table A6.6: Polychlorinated dibenzofuranes (PCDFs) (...continued)

| Substance<br>Formula<br>(Trivial Name)<br>[CAS Registry Number]<br>InChIKey | $H_s^{cp}$<br>(at $T^\ominus$)<br>$\left[\dfrac{\text{mol}}{\text{m}^3\,\text{Pa}}\right]$ | $\dfrac{\text{d}\ln H_s^{cp}}{\text{d}(1/T)}$<br><br>[K] | Reference | Type | Note |
|---|---|---|---|---|---|
| 1,2,3,4,7,9-<br>hexachlorodibenzofuran<br>$C_{12}H_2Cl_6O$<br>(PCDF-123479)<br>[91538-84-0]<br>BKIXWRBZCQEZAQ-UHFFFAOYSA-N | 1.1 | | Govers and Krop (1998) | Q | |
| 1,2,3,4,8,9-<br>hexachlorodibenzofuran<br>$C_{12}H_2Cl_6O$<br>(PCDF-123489)<br>[92341-07-6]<br>VSDUQUBUJRNREH-UHFFFAOYSA-N | 2.7<br><br>1.1 | | Saçan et al. (2005)<br><br>Govers and Krop (1998) | Q<br><br>Q | |
| 1,2,3,6,7,8-<br>hexachlorodibenzofuran<br>$C_{12}H_2Cl_6O$<br>(PCDF-123678)<br>[57117-44-9]<br>JEYJJJXOFWNEHN-UHFFFAOYSA-N | $9.1\times10^{-1}$<br><br>$1.1\times10^{-3}$<br>2.2<br>$5.2\times10^{-1}$ | <br><br>3300 | Govers and Krop (1998)<br><br>Paasivirta et al. (1999)<br>Saçan et al. (2005)<br>Govers and Krop (1998) | V<br><br>T<br>Q<br>Q | |
| 1,2,3,6,7,9-<br>hexachlorodibenzofuran<br>$C_{12}H_2Cl_6O$<br>(PCDF-123679)<br>[92341-06-5]<br>JZVOLXQREJNTTL-UHFFFAOYSA-N | 1.0 | | Govers and Krop (1998) | Q | |
| 1,2,3,6,8,9-<br>hexachlorodibenzofuran<br>$C_{12}H_2Cl_6O$<br>(PCDF-123689)<br>[75198-38-8]<br>WLGQZUOHEXTWFH-UHFFFAOYSA-N | 1.3 | | Govers and Krop (1998) | Q | |
| 1,2,3,7,8,9-<br>hexachlorodibenzofuran<br>$C_{12}H_2Cl_6O$<br>(PCDF-123789)<br>[72918-21-9]<br>PYUSJFJVDVSXIU-UHFFFAOYSA-N | $6.3\times10^{-4}$<br><br>2.6<br>1.0 | 2600 | Paasivirta et al. (1999)<br><br>Saçan et al. (2005)<br>Govers and Krop (1998) | T<br><br>Q<br>Q | |
| 1,2,4,6,7,8-<br>hexachlorodibenzofuran<br>$C_{12}H_2Cl_6O$<br>(PCDF-124678)<br>[67562-40-7]<br>CKDDYGBBKXIHRY-UHFFFAOYSA-N | $3.2\times10^{-4}$<br><br>$9.3\times10^{-1}$ | 2300 | Paasivirta et al. (1999)<br><br>Govers and Krop (1998) | T<br><br>Q | |



Table A6.6: Polychlorinated dibenzofuranes (PCDFs) (...continued)

| Substance<br>Formula<br>(Trivial Name)<br>[CAS Registry Number]<br>InChIKey | $H_s^{cp}$<br>(at $T^\ominus$)<br>$\left[\dfrac{\mathrm{mol}}{\mathrm{m^3\,Pa}}\right]$ | $\dfrac{\mathrm{d}\ln H_s^{cp}}{\mathrm{d}(1/T)}$<br><br>[K] | Reference | Type | Note |
|---|---|---|---|---|---|
| 1,2,4,6,7,9-<br>hexachlorodibenzofuran<br>$C_{12}H_2Cl_6O$<br>(PCDF-124679)<br>[75627-02-0]<br>FAHIPPHRJJJQOG-UHFFFAOYSA-N | 1.5 | | Govers and Krop (1998) | Q | |
| 1,2,4,6,8,9-<br>hexachlorodibenzofuran<br>$C_{12}H_2Cl_6O$<br>(PCDF-124689)<br>[69698-59-5]<br>FGZCXIPKUCJGHW-UHFFFAOYSA-N | $2.2\times10^{-4}$<br><br>2.4 | 2600 | Paasivirta et al. (1999)<br><br>Govers and Krop (1998) | T<br><br>Q | |
| 1,3,4,6,7,8-<br>hexachlorodibenzofuran<br>$C_{12}H_2Cl_6O$<br>(PCDF-134678)<br>[71998-75-9]<br>OGTZINCGZYXZCR-UHFFFAOYSA-N | 1.1 | | Govers and Krop (1998) | Q | |
| 1,3,4,6,7,9-<br>hexachlorodibenzofuran<br>$C_{12}H_2Cl_6O$<br>(PCDF-134679)<br>[92341-05-4]<br>TURXXOIFVABBPW-UHFFFAOYSA-N | 1.6 | | Govers and Krop (1998) | Q | |
| 2,3,4,6,7,8-<br>hexachlorodibenzofuran<br>$C_{12}H_2Cl_6O$<br>(PCDF-234678)<br>[60851-34-5]<br>XTAHLACQOVXINQ-UHFFFAOYSA-N | $3.6\times10^{-4}$<br><br>3.1<br>$5.6\times10^{-1}$ | 2600 | Paasivirta et al. (1999)<br><br>Saçan et al. (2005)<br>Govers and Krop (1998) | T<br><br>Q<br>Q | |
| 1,2,3,4,6,7,8-<br>heptachlorodibenzofuran<br>$C_{12}HCl_7O$<br>(PCDF-1234678)<br>[67562-39-4]<br>WDMKCPIVJOGHBF-UHFFFAOYSA-N | $7.0\times10^{-1}$<br><br>$7.0\times10^{-1}$<br>$2.9\times10^{-1}$<br>$5.4\times10^{-5}$<br>2.1<br>3.9<br>$7.1\times10^{-1}$ | <br><br><br><br>1600 | Duchowicz et al. (2020)<br><br>Mackay et al. (2006b)<br>Govers and Krop (1998)<br>Paasivirta et al. (1999)<br>Duchowicz et al. (2020)<br>Saçan et al. (2005)<br>Govers and Krop (1998) | V<br><br>V<br>V<br>T<br>Q<br>Q<br>Q | 186 |
| 1,2,3,4,6,7,9-<br>heptachlorodibenzofuran<br>$C_{12}HCl_7O$<br>(PCDF-1234679)<br>[70648-25-8]<br>JWXWOEUKHXOUSK-UHFFFAOYSA-N | 1.5 | | Govers and Krop (1998) | Q | |



Table A6.6: Polychlorinated dibenzofuranes (PCDFs) (...continued)

| Substance Formula (Trivial Name) [CAS Registry Number] InChIKey | $H_s^{cp}$ (at $T^{\ominus}$) $\left[\dfrac{\text{mol}}{\text{m}^3\,\text{Pa}}\right]$ | $\dfrac{\text{d}\ln H_s^{cp}}{\text{d}(1/T)}$ [K] | Reference | Type | Note |
|---|---|---|---|---|---|
| 1,2,3,4,6,8,9-heptachlorodibenzofuran $C_{12}HCl_7O$ (PCDF-1234689) [69698-58-4] BADFHCOLISGRRW-UHFFFAOYSA-N | $3.4\times10^{-4}$ <br> 1.9 | 1800 | Paasivirta et al. (1999) <br> Govers and Krop (1998) | T <br> Q | |
| 1,2,3,4,7,8,9-heptachlorodibenzofuran $C_{12}HCl_7O$ (PCDF-1234789) [55673-89-7] VEZCTZWLJYWARH-UHFFFAOYSA-N | $5.5\times10^{-4}$ <br> 3.2 <br> 1.0 | 2100 | Paasivirta et al. (1999) <br> Saçan et al. (2005) <br> Govers and Krop (1998) | T <br> Q <br> Q | |
| octachlorodibenzofuran $C_{12}Cl_8O$ (PCDF-12346789) [39001-02-0] RHIROFAGUQOFLU-UHFFFAOYSA-N | 5.2 <br> <br> $7.6\times10^{-1}$ <br> $2.3\times10^{-4}$ <br> 1.7 <br> 4.9 <br> 1.3 | 2400 | Duchowicz et al. (2020) <br> Mackay et al. (2006b) <br> Govers and Krop (1998) <br> Paasivirta et al. (1999) <br> Duchowicz et al. (2020) <br> Saçan et al. (2005) <br> Govers and Krop (1998) | V <br> V <br> V <br> T <br> Q <br> Q <br> Q | 186 <br> 683 |



### A6.7 Polychlorinated dibenzo-$p$-dioxins (PCDDs)

Table A6.7: Polychlorinated dibenzo-$p$-dioxins (PCDDs)

| Substance Formula (Trivial Name) [CAS Registry Number] InChIKey | $H_s^{cp}$ (at $T^{\ominus}$) $\left[\dfrac{\text{mol}}{\text{m}^3\,\text{Pa}}\right]$ | $\dfrac{\mathrm{d}\ln H_s^{cp}}{\mathrm{d}(1/T)}$ [K] | Reference | Type | Note |
|---|---|---|---|---|---|
| 1-chlorodibenzo-$p$-dioxin $C_{12}H_7ClO_2$ (PCDD-1) [39227-53-7] VGGGRWRBGXENKI-UHFFFAOYSA-N | $1.6\times10^{-1}$ | | Duchowicz et al. (2020) | V | 186 |
| | $1.6\times10^{-1}$ | | Mackay et al. (2006b) | V | |
| | $2.5\times10^{-2}$ | | Saçan et al. (2005) | V | |
| | $1.6\times10^{-1}$ | | Govers and Krop (1998) | V | |
| | $1.2\times10^{-1}$ | | Shiu et al. (1988) | V | |
| | $2.1$ | | Duchowicz et al. (2020) | Q | |
| | | $7100$ | Kühne et al. (2005) | Q | |
| | $6.8\times10^{-2}$ | | Saçan et al. (2005) | Q | |
| | $1.3\times10^{-1}$ | | Wang and Wong (2002) | Q | 535 |
| | $1.7\times10^{-1}$ | | Govers and Krop (1998) | Q | |
| | | $6500$ | Kühne et al. (2005) | ? | |
| 2-chlorodibenzo-$p$-dioxin $C_{12}H_7ClO_2$ (PCDD-2) [39227-54-8] GIUGGRUEPHPVNR-UHFFFAOYSA-N | $8.6\times10^{-1}$ | | Duchowicz et al. (2020) | V | 186 |
| | $7.9\times10^{-2}$ | | Mackay et al. (2006b) | V | |
| | $7.9\times10^{-2}$ | | Govers and Krop (1998) | V | |
| | $6.7\times10^{-2}$ | | Shiu et al. (1988) | V | |
| | $1.1$ | | Duchowicz et al. (2020) | Q | |
| | $9.8\times10^{-2}$ | | Saçan et al. (2005) | Q | |
| | $1.3\times10^{-1}$ | | Wang and Wong (2002) | Q | 535 |
| | $2.2\times10^{-1}$ | | Govers and Krop (1998) | Q | |
| 1,2-dichlorodibenzo-$p$-dioxin $C_{12}H_6Cl_2O_2$ (PCDD-12) [54536-18-4] DFGDMWHUCCHXIF-UHFFFAOYSA-N | $2.8\times10^{-1}$ | | Wang and Wong (2002) | Q | 535 |
| | $3.2\times10^{-1}$ | | Govers and Krop (1998) | Q | |
| 1,3-dichlorodibenzo-$p$-dioxin $C_{12}H_6Cl_2O_2$ (PCDD-13) [50585-39-2] AZYJYMAKTBXNSX-UHFFFAOYSA-N | $2.2\times10^{-1}$ | | Wang and Wong (2002) | Q | 535 |
| | $3.8\times10^{-1}$ | | Govers and Krop (1998) | Q | |
| 1,4-dichlorodibenzo-$p$-dioxin $C_{12}H_6Cl_2O_2$ (PCDD-14) [54536-19-5] MBMUPQZSDWVPQU-UHFFFAOYSA-N | $2.4\times10^{-1}$ | | Wang and Wong (2002) | Q | 535 |
| | $3.2\times10^{-1}$ | | Govers and Krop (1998) | Q | |
| 1,6-dichlorodibenzo-$p$-dioxin $C_{12}H_6Cl_2O_2$ (PCDD-16) [38178-38-0] MAWMBEVNJGEDAD-UHFFFAOYSA-N | $2.5\times10^{-1}$ | | Wang and Wong (2002) | Q | 535 |
| | $3.2\times10^{-1}$ | | Govers and Krop (1998) | Q | |



Table A6.7: Polychlorinated dibenzo-$p$-dioxins (PCDDs) (. . . continued)

| Substance Formula (Trivial Name) [CAS Registry Number] InChIKey | $H_s^{cp}$ (at $T^\ominus$) $\left[\dfrac{\text{mol}}{\text{m}^3\,\text{Pa}}\right]$ | $\dfrac{\text{d}\ln H_s^{cp}}{\text{d}(1/T)}$ [K] | Reference | Type | Note |
|---|---|---|---|---|---|
| 1,7-dichlorodibenzo-$p$-dioxin $C_{12}H_6Cl_2O_2$ (PCDD-17) [82291-26-7] IJUWLAFPPVRYGY-UHFFFAOYSA-N | $2.6\times10^{-1}$ $3.6\times10^{-1}$ | | Wang and Wong (2002) Govers and Krop (1998) | Q Q | 535 |
| 1,8-dichlorodibenzo-$p$-dioxin $C_{12}H_6Cl_2O_2$ (PCDD-18) [82291-27-8] PLZYIHQBHROTFD-UHFFFAOYSA-N | $2.6\times10^{-1}$ $3.8\times10^{-1}$ | | Wang and Wong (2002) Govers and Krop (1998) | Q Q | 535 |
| 1,9-dichlorodibenzo-$p$-dioxin $C_{12}H_6Cl_2O_2$ (PCDD-19) [82291-28-9] JZDVJXBKJDADAY-UHFFFAOYSA-N | $2.6\times10^{-1}$ $5.4\times10^{-1}$ | | Wang and Wong (2002) Govers and Krop (1998) | Q Q | 535 |
| 2,3-dichlorodibenzo-$p$-dioxin $C_{12}H_6Cl_2O_2$ (PCDD-23) [29446-15-9] YCIYTXRUZSDMRZ-UHFFFAOYSA-N | $1.5\times10^{-1}$ $1.5\times10^{-1}$ $1.5\times10^{-1}$ $1.5\times10^{-1}$ $1.5\times10^{-1}$ $8.5\times10^{-1}$ $2.5\times10^{-1}$ $2.6\times10^{-1}$ $4.0\times10^{-1}$ | | Duchowicz et al. (2020) Mackay et al. (2006b) Saçan et al. (2005) Govers and Krop (1998) Shiu et al. (1988) Duchowicz et al. (2020) Saçan et al. (2005) Wang and Wong (2002) Govers and Krop (1998) | V V V V V Q Q Q Q | 186 535 |
| 2,7-dichlorodibenzo-$p$-dioxin $C_{12}H_6Cl_2O_2$ (PCDD-27) [33857-26-0] NBFMTHWVRBOVPE-UHFFFAOYSA-N | $1.7\times10^{-1}$ $1.2\times10^{-1}$ $1.2\times10^{-1}$ $2.4\times10^{-1}$ $6.6\times10^{-1}$ $7.3\times10^{-1}$ $5.0$ $1.0\times10^{-1}$ $2.6\times10^{-1}$ $3.5\times10^{-1}$ $1.7\times10^{-1}$ | | Santl et al. (1994) Mackay et al. (2006b) Govers and Krop (1998) Shiu et al. (1988) Keshavarz et al. (2022) Duchowicz et al. (2020) Hilal et al. (2008) Modarresi et al. (2007) Saçan et al. (2005) Wang and Wong (2002) Govers and Krop (1998) Duchowicz et al. (2020) | M V V V Q Q Q Q Q Q Q ? | 733 683 299 67 535 185, 21 |
| 2,8-dichlorodibenzo-$p$-dioxin $C_{12}H_6Cl_2O_2$ (PCDD-28) [38964-22-6] WMWJCKBJUQDYLM-UHFFFAOYSA-N | $4.7\times10^{-1}$ $4.7\times10^{-1}$ $4.7\times10^{-1}$ $4.7\times10^{-1}$ $1.2$ $1.7\times10^{-1}$ $2.6\times10^{-1}$ $4.4\times10^{-1}$ | | Duchowicz et al. (2020) Mackay et al. (2006b) Govers and Krop (1998) Shiu et al. (1988) Duchowicz et al. (2020) Saçan et al. (2005) Wang and Wong (2002) Govers and Krop (1998) | V V V V Q Q Q Q | 186 535 |



Table A6.7: Polychlorinated dibenzo-$p$-dioxins (PCDDs) (. . . continued)

| Substance Formula (Trivial Name) [CAS Registry Number] InChIKey | $H_s^{cp}$ (at $T^{\ominus}$) $\left[\dfrac{\mathrm{mol}}{\mathrm{m^3\,Pa}}\right]$ | $\dfrac{\mathrm{d}\ln H_s^{cp}}{\mathrm{d}(1/T)}$ [K] | Reference | Type | Note |
|---|---|---|---|---|---|
| 1,2,3-trichlorodibenzo-$p$-dioxin $C_{12}H_5Cl_3O_2$ (PCDD-123) [54536-17-3] SKMFBGZVVNDVFR-UHFFFAOYSA-N | $5.0\times10^{-1}$ $5.6\times10^{-1}$ | | Wang and Wong (2002) Govers and Krop (1998) | Q Q | 535 |
| 1,2,4-trichlorodibenzo-$p$-dioxin $C_{12}H_5Cl_3O_2$ (PCDD-124) [39227-58-2] HRVUKLBFRPWXPJ-UHFFFAOYSA-N | $2.7\times10^{-1}$ $2.9\times10^{-1}$ $2.6\times10^{-1}$ $2.6\times10^{-1}$ $2.4\times10^{-1}$ $9.1\times10^{-1}$ 1.3 $1.1\times10^{1}$ $3.0\times10^{-1}$ $4.4\times10^{-1}$ $5.5\times10^{-1}$ $2.7\times10^{-1}$ | | Santl et al. (1994) Mackay et al. (2006b) Govers and Krop (1998) Shiu et al. (1988) Keshavarz et al. (2022) Duchowicz et al. (2020) Hilal et al. (2008) Modarresi et al. (2007) Saçan et al. (2005) Wang and Wong (2002) Govers and Krop (1998) Duchowicz et al. (2020) Fogg and Sangster (2003) | M V V V Q Q Q Q Q Q Q ? W | 733 67 535 185, 21 734 |
| 1,2,6-trichlorodibenzo-$p$-dioxin $C_{12}H_5Cl_3O_2$ (PCDD-126) [69760-96-9] XQBPVWBIUBCJJO-UHFFFAOYSA-N | $5.0\times10^{-1}$ $5.1\times10^{-1}$ | | Wang and Wong (2002) Govers and Krop (1998) | Q Q | 535 |
| 1,2,7-trichlorodibenzo-$p$-dioxin $C_{12}H_5Cl_3O_2$ (PCDD-127) [82291-30-3] TXJMXDWFPQSYEQ-UHFFFAOYSA-N | $5.1\times10^{-1}$ $4.6\times10^{-1}$ | | Wang and Wong (2002) Govers and Krop (1998) | Q Q | 535 |
| 1,2,8-trichlorodibenzo-$p$-dioxin $C_{12}H_5Cl_3O_2$ (PCDD-128) [82291-31-4] QBEOCKSANJLBAE-UHFFFAOYSA-N | $5.1\times10^{-1}$ $6.0\times10^{-1}$ | | Wang and Wong (2002) Govers and Krop (1998) | Q Q | 535 |
| 1,2,9-trichlorodibenzo-$p$-dioxin $C_{12}H_5Cl_3O_2$ (PCDD-129) [82291-32-5] DQLRDBDQLSIOIX-UHFFFAOYSA-N | $5.2\times10^{-1}$ $9.1\times10^{-1}$ | | Wang and Wong (2002) Govers and Krop (1998) | Q Q | 535 |
| 1,3,6-trichlorodibenzo-$p$-dioxin $C_{12}H_5Cl_3O_2$ (PCDD-136) [82291-33-6] LNPVMVSAUXUGHH-UHFFFAOYSA-N | $4.2\times10^{-1}$ $6.3\times10^{-1}$ | | Wang and Wong (2002) Govers and Krop (1998) | Q Q | 535 |



Table A6.7: Polychlorinated dibenzo-$p$-dioxins (PCDDs) (...continued)

| Substance Formula (Trivial Name) [CAS Registry Number] InChIKey | $H_s^{cp}$ (at $T^{\ominus}$) $\left[\dfrac{\mathrm{mol}}{\mathrm{m^3\,Pa}}\right]$ | $\dfrac{\mathrm{d}\ln H_s^{cp}}{\mathrm{d}(1/T)}$ [K] | Reference | Type | Note |
|---|---|---|---|---|---|
| 1,3,7-trichlorodibenzo-$p$-dioxin $C_{12}H_5Cl_3O_2$ (PCDD-137) [67028-17-5] RPKWIXFZKMDPMH-UHFFFAOYSA-N | $4.3\times10^{-1}$ $6.8\times10^{-1}$ | | Wang and Wong (2002) Govers and Krop (1998) | Q Q | 535 |
| 1,3,8-trichlorodibenzo-$p$-dioxin $C_{12}H_5Cl_3O_2$ (PCDD-138) [82306-61-4] FJAKCOBYQSEWMT-UHFFFAOYSA-N | $4.3\times10^{-1}$ $5.6\times10^{-1}$ | | Wang and Wong (2002) Govers and Krop (1998) | Q Q | 535 |
| 1,3,9-trichlorodibenzo-$p$-dioxin $C_{12}H_5Cl_3O_2$ (PCDD-139) [82306-62-5] DGDADRUTFAIIQQ-UHFFFAOYSA-N | $4.4\times10^{-1}$ $1.0$ | | Wang and Wong (2002) Govers and Krop (1998) | Q Q | 535 |
| 1,4,6-trichlorodibenzo-$p$-dioxin $C_{12}H_5Cl_3O_2$ (PCDD-146) [82306-63-6] UTTYFTWIJLRXKB-UHFFFAOYSA-N | $4.4\times10^{-1}$ $9.3\times10^{-1}$ | | Wang and Wong (2002) Govers and Krop (1998) | Q Q | 535 |
| 1,4,7-trichlorodibenzo-$p$-dioxin $C_{12}H_5Cl_3O_2$ (PCDD-147) [82306-64-7] NBWAQBGJBSYXHV-UHFFFAOYSA-N | $4.5\times10^{-1}$ $6.0\times10^{-1}$ | | Wang and Wong (2002) Govers and Krop (1998) | Q Q | 535 |
| 1,7,8-trichlorodibenzo-$p$-dioxin $C_{12}H_5Cl_3O_2$ (PCDD-178) [82306-65-8] CAPCTZJHYADFNX-UHFFFAOYSA-N | $4.9\times10^{-1}$ $6.2\times10^{-1}$ | | Wang and Wong (2002) Govers and Krop (1998) | Q Q | 535 |
| 2,3,7-trichlorodibenzo-$p$-dioxin $C_{12}H_5Cl_3O_2$ (PCDD-237) [33857-28-2] ZSIZNEVHVVRPFF-UHFFFAOYSA-N | $4.9\times10^{-1}$ $5.6\times10^{-1}$ | | Wang and Wong (2002) Govers and Krop (1998) | Q Q | 535 |
| 1,2,3,4-tetrachlorodibenzo-$p$-dioxin $C_{12}H_4Cl_4O_2$ (PCDD-1234) [30746-58-8] DJHHDLMTUOLVHY-UHFFFAOYSA-N | $5.0\times10^{-1}$ $2.7\times10^{-1}$ $1.4$ $3.3\times10^{-1}$ $2.7\times10^{-1}$ $2.4\times10^{-1}$ $7.0\times10^{-1}$ $1.4$ | | Santl et al. (1994) Mackay et al. (2006b) Mackay et al. (2006b) Govers and Krop (1998) Shiu et al. (1988) Keshavarz et al. (2022) Duchowicz et al. (2020) Hilal et al. (2008) | M V V V V Q Q Q | 733 184 |



Table A6.7: Polychlorinated dibenzo-$p$-dioxins (PCDDs) (...continued)

| Substance Formula (Trivial Name) [CAS Registry Number] InChIKey | $H_s^{cp}$ (at $T^\ominus$) $\left[\dfrac{\mathrm{mol}}{\mathrm{m^3\,Pa}}\right]$ | $\dfrac{\mathrm{d}\ln H_s^{cp}}{\mathrm{d}(1/T)}$ [K] | Reference | Type | Note |
|---|---|---|---|---|---|
| | $2.2\times10^1$ | | Modarresi et al. (2007) | Q | 67 |
| | $6.3\times10^{-1}$ | | Saçan et al. (2005) | Q | |
| | $8.7\times10^{-1}$ | | Wang and Wong (2002) | Q | 535 |
| | $7.4\times10^{-1}$ | | Govers and Krop (1998) | Q | |
| | $4.9\times10^{-1}$ | | Duchowicz et al. (2020) | ? | 185, 21 |
| 1,2,3,6-tetrachlorodibenzo-$p$-dioxin | $8.7\times10^{-1}$ | | Wang and Wong (2002) | Q | 535 |
| $C_{12}H_4Cl_4O_2$ (PCDD-1236) [71669-25-5] XEZBZSVTUSXISZ-UHFFFAOYSA-N | $8.1\times10^{-1}$ | | Govers and Krop (1998) | Q | |
| 1,2,3,7-tetrachlorodibenzo-$p$-dioxin | 1.3 | | Mackay et al. (2006b) | V | |
| $C_{12}H_4Cl_4O_2$ | 1.7 | | Govers and Krop (1998) | V | |
| (PCDD-1237) | 1.3 | | Shiu et al. (1988) | V | |
| [67028-18-6] | $2.4\times10^{-1}$ | | Keshavarz et al. (2022) | Q | |
| SKGXYFVQZVPEFP-UHFFFAOYSA-N | $5.7\times10^{-1}$ | | Duchowicz et al. (2020) | Q | 184 |
| | 1.7 | | Hilal et al. (2008) | Q | |
| | $2.1\times10^1$ | | Modarresi et al. (2007) | Q | 67 |
| | $4.3\times10^{-1}$ | | Saçan et al. (2005) | Q | |
| | $8.7\times10^{-1}$ | | Wang and Wong (2002) | Q | 535 |
| | $6.8\times10^{-1}$ | | Govers and Krop (1998) | Q | |
| | 1.3 | | Duchowicz et al. (2020) | ? | 185, 21 |
| 1,2,3,8-tetrachlorodibenzo-$p$-dioxin | $8.7\times10^{-1}$ | | Wang and Wong (2002) | Q | 535 |
| $C_{12}H_4Cl_4O_2$ (PCDD-1238) [53555-02-5] BXKLTNKYLCZOHF-UHFFFAOYSA-N | $7.1\times10^{-1}$ | | Govers and Krop (1998) | Q | |
| 1,2,3,9-tetrachlorodibenzo-$p$-dioxin | $9.1\times10^{-1}$ | | Wang and Wong (2002) | Q | 535 |
| $C_{12}H_4Cl_4O_2$ (PCDD-1239) [71669-26-6] CMVHZKSHSHQJHS-UHFFFAOYSA-N | 1.4 | | Govers and Krop (1998) | Q | |
| 1,2,4,6-tetrachlorodibenzo-$p$-dioxin | $8.1\times10^{-1}$ | | Wang and Wong (2002) | Q | 535 |
| $C_{12}H_4Cl_4O_2$ (PCDD-1246) [71669-27-7] KQNBZUDHTCXCNA-UHFFFAOYSA-N | 1.4 | | Govers and Krop (1998) | Q | |





Table A6.7: Polychlorinated dibenzo-$p$-dioxins (PCDDs) (. . . continued)

| Substance Formula (Trivial Name) [CAS Registry Number] InChIKey | $H_s^{cp}$ (at $T^{\ominus}$) $\left[\dfrac{\text{mol}}{\text{m}^3\,\text{Pa}}\right]$ | $\dfrac{\text{d}\ln H_s^{cp}}{\text{d}(1/T)}$ [K] | Reference | Type | Note |
|---|---|---|---|---|---|
| 1,2,4,7-tetrachlorodibenzo-$p$-dioxin | $7.8\times10^{-1}$ | | Wang and Wong (2002) | Q | 535 |
| C$_{12}$H$_4$Cl$_4$O$_2$ | $7.1\times10^{-1}$ | | Govers and Krop (1998) | Q | |
| (PCDD-1247) | | | | | |
| [71669-28-8] | | | | | |
| SMPHQCMJQUBTFZ-UHFFFAOYSA-N | | | | | |
| 1,2,4,8-tetrachlorodibenzo-$p$-dioxin | $7.8\times10^{-1}$ | | Wang and Wong (2002) | Q | 535 |
| C$_{12}$H$_4$Cl$_4$O$_2$ | $8.9\times10^{-1}$ | | Govers and Krop (1998) | Q | |
| (PCDD-1248) | | | | | |
| [71669-29-9] | | | | | |
| XGIKODBWQSAEFQ-UHFFFAOYSA-N | | | | | |
| 1,2,4,9-tetrachlorodibenzo-$p$-dioxin | $8.1\times10^{-1}$ | | Wang and Wong (2002) | Q | 535 |
| C$_{12}$H$_4$Cl$_4$O$_2$ | 1.5 | | Govers and Krop (1998) | Q | |
| (PCDD-1249) | | | | | |
| [71665-99-1] | | | | | |
| WDAHVJCSSYOALR-UHFFFAOYSA-N | | | | | |
| 1,2,6,7-tetrachlorodibenzo-$p$-dioxin | $9.5\times10^{-1}$ | | Wang and Wong (2002) | Q | 535 |
| C$_{12}$H$_4$Cl$_4$O$_2$ | $5.8\times10^{-1}$ | | Govers and Krop (1998) | Q | |
| (PCDD-1267) | | | | | |
| [40581-90-6] | | | | | |
| SAMLAWFHXZIRMP-UHFFFAOYSA-N | | | | | |
| 1,2,6,8-tetrachlorodibenzo-$p$-dioxin | $8.1\times10^{-1}$ | | Wang and Wong (2002) | Q | 535 |
| C$_{12}$H$_4$Cl$_4$O$_2$ | $8.9\times10^{-1}$ | | Govers and Krop (1998) | Q | |
| (PCDD-1268) | | | | | |
| [67323-56-2] | | | | | |
| YYUFYZDSYHKVDP-UHFFFAOYSA-N | | | | | |
| 1,2,6,9-tetrachlorodibenzo-$p$-dioxin | $8.7\times10^{-1}$ | | Wang and Wong (2002) | Q | 535 |
| C$_{12}$H$_4$Cl$_4$O$_2$ | 1.4 | | Govers and Krop (1998) | Q | |
| (PCDD-1269) | | | | | |
| [40581-91-7] | | | | | |
| ZKMXKYXNLFLUCD-UHFFFAOYSA-N | | | | | |
| 1,2,7,8-tetrachlorodibenzo-$p$-dioxin | $7.8\times10^{-1}$ | | Wang and Wong (2002) | Q | 535 |
| C$_{12}$H$_4$Cl$_4$O$_2$ | $6.8\times10^{-1}$ | | Govers and Krop (1998) | Q | |
| (PCDD-1278) | | | | | |
| [34816-53-0] | | | | | |
| YDZCLBKUTXYYKS-UHFFFAOYSA-N | | | | | |





Table A6.7: Polychlorinated dibenzo-$p$-dioxins (PCDDs) (...continued)

| Substance Formula (Trivial Name) [CAS Registry Number] InChIKey | $H_s^{cp}$ (at $T^\ominus$) $\left[\dfrac{\mathrm{mol}}{\mathrm{m^3\,Pa}}\right]$ | $\dfrac{\mathrm{d}\ln H_s^{cp}}{\mathrm{d}(1/T)}$ [K] | Reference | Type | Note |
|---|---|---|---|---|---|
| 1,2,7,9-tetrachlorodibenzo-$p$-dioxin | $8.3\times10^{-1}$ | | Wang and Wong (2002) | Q | 535 |
| C$_{12}$H$_4$Cl$_4$O$_2$ | 1.2 | | Govers and Krop (1998) | Q | |
| (PCDD-1279) | | | | | |
| [71669-23-3] | | | | | |
| QIKHBBZEUNSCAF-UHFFFAOYSA-N | | | | | |
| 1,2,8,9-tetrachlorodibenzo-$p$-dioxin | $9.8\times10^{-1}$ | | Wang and Wong (2002) | Q | 535 |
| C$_{12}$H$_4$Cl$_4$O$_2$ | 1.3 | | Govers and Krop (1998) | Q | |
| (PCDD-1289) | | | | | |
| [62470-54-6] | | | | | |
| WELWFAGPAZKSBG-UHFFFAOYSA-N | | | | | |
| 1,3,6,8-tetrachlorodibenzo-$p$-dioxin | $1.4\times10^{-1}$ | | Webster et al. (1985) | M | |
| C$_{12}$H$_4$Cl$_4$O$_2$ | 1.4 | | Govers and Krop (1998) | V | |
| (PCDD-1368) | 1.4 | | Shiu et al. (1988) | V | |
| [33423-92-6] | 1.2 | | Hilal et al. (2008) | Q | |
| OTQFXRBLGNEOGH-UHFFFAOYSA-N | $2.9\times10^{-1}$ | | Saçan et al. (2005) | Q | |
| | $6.8\times10^{-1}$ | | Wang and Wong (2002) | Q | 535 |
| | $8.7\times10^{-1}$ | | Govers and Krop (1998) | Q | |
| 1,3,6,9-tetrachlorodibenzo-$p$-dioxin | $7.4\times10^{-1}$ | | Wang and Wong (2002) | Q | 535 |
| C$_{12}$H$_4$Cl$_4$O$_2$ | 1.7 | | Govers and Krop (1998) | Q | |
| (PCDD-1369) | | | | | |
| [71669-24-4] | | | | | |
| QAUIRDIJIUMMEP-UHFFFAOYSA-N | | | | | |
| 1,3,7,8-tetrachlorodibenzo-$p$-dioxin | $7.8\times10^{-1}$ | | Wang and Wong (2002) | Q | 535 |
| C$_{12}$H$_4$Cl$_4$O$_2$ | $7.9\times10^{-1}$ | | Govers and Krop (1998) | Q | |
| (PCDD-1378) | | | | | |
| [50585-46-1] | | | | | |
| VPTDIAYLYJBYQG-UHFFFAOYSA-N | | | | | |
| 1,3,7,9-tetrachlorodibenzo-$p$-dioxin | $7.1\times10^{-1}$ | | Wang and Wong (2002) | Q | 535 |
| C$_{12}$H$_4$Cl$_4$O$_2$ | 1.7 | | Govers and Krop (1998) | Q | |
| (PCDD-1379) | | | | | |
| [62470-53-5] | | | | | |
| JMGYHLJVDHUACM-UHFFFAOYSA-N | | | | | |
| 1,4,6,9-tetrachlorodibenzo-$p$-dioxin | $7.9\times10^{-1}$ | | Wang and Wong (2002) | Q | 535 |
| C$_{12}$H$_4$Cl$_4$O$_2$ | 2.6 | | Govers and Krop (1998) | Q | |
| (PCDD-1469) | | | | | |
| [40581-93-9] | | | | | |
| QTIIAIRUSSSOHT-UHFFFAOYSA-N | | | | | |



Table A6.7: Polychlorinated dibenzo-$p$-dioxins (PCDDs) (. . . continued)

| Substance<br>Formula<br>(Trivial Name)<br>[CAS Registry Number]<br>InChIKey | $H_s^{cp}$<br>(at $T^{\ominus}$)<br>$\left[\dfrac{\mathrm{mol}}{\mathrm{m^3\,Pa}}\right]$ | $\dfrac{\mathrm{d}\ln H_s^{cp}}{\mathrm{d}(1/T)}$<br><br>[K] | Reference | Type | Note |
|---|---|---|---|---|---|
| 1,4,7,8-tetrachlorodibenzo-$p$-dioxin<br>$C_{12}H_4Cl_4O_2$<br>(PCDD-1478)<br>[40581-94-0]<br>FCRXUTCUWCJZJI-UHFFFAOYSA-N | $8.1\times10^{-1}$<br><br>$9.1\times10^{-1}$ | | Wang and Wong (2002)<br><br>Govers and Krop (1998) | Q<br><br>Q | 535 |
| 2,3,7,8-tetrachlorodibenzo-$p$-dioxin<br>$C_{12}H_4Cl_4O_2$<br>(PCDD-2378; TCDD)<br>[1746-01-6]<br>HGUFODBRKLSHSI-UHFFFAOYSA-N | $2.0\times10^{-1}$<br>$2.0\times10^{-1}$<br>$3.0\times10^{-1}$<br>$3.0\times10^{-1}$<br>$5.8\times10^{-1}$<br>$6.1\times10^{-1}$<br>$3.0\times10^{-1}$<br>$9.7\times10^{-2}$<br>$6.3\times10^{-1}$<br>4.7<br>$2.6\times10^{-4}$<br>$4.1\times10^{-1}$<br>$3.3\times10^{-1}$<br>$8.9\times10^{-1}$<br>$6.2\times10^{-1}$ | <br><br><br><br><br><br><br><br><br><br>3600 | Duchowicz et al. (2020)<br>HSDB (2015)<br>Mackay et al. (2006b)<br>Govers and Krop (1998)<br>McLachlan et al. (1990)<br>Shiu et al. (1988)<br>Shiu et al. (1988)<br>Shiu et al. (1988)<br>Podoll et al. (1986)<br>Schroy et al. (1985)<br>Paasivirta et al. (1999)<br>Duchowicz et al. (2020)<br>Saçan et al. (2005)<br>Wang and Wong (2002)<br>Govers and Krop (1998) | V<br>V<br>V<br>V<br>V<br>V<br>V<br>V<br>V<br>V<br>T<br>Q<br>Q<br>Q<br>Q | 186<br><br><br><br>373<br><br><br><br><br><br><br><br><br>535 |
| 1,2,3,4,6-pentachlorodibenzo-$p$-dioxin<br>$C_{12}H_3Cl_5O_2$<br>(PCDD-12346)<br>[67028-19-7]<br>LNWDBNKKBLRAMH-UHFFFAOYSA-N | 1.5<br><br>1.8 | | Wang and Wong (2002)<br><br>Govers and Krop (1998) | Q<br><br>Q | 535 |
| 1,2,3,4,7-pentachlorodibenzo-$p$-dioxin<br>$C_{12}H_3Cl_5O_2$<br>(PCDD-12347)<br>[39227-61-7]<br>WRNGAZFESPEMCN-UHFFFAOYSA-N | 3.8<br>3.8<br>4.5<br>3.8<br>$4.5\times10^{-1}$<br>$7.0\times10^{-1}$<br>1.4<br>$8.1\times10^{-1}$ | | Duchowicz et al. (2020)<br>Mackay et al. (2006b)<br>Govers and Krop (1998)<br>Shiu et al. (1988)<br>Duchowicz et al. (2020)<br>Saçan et al. (2005)<br>Wang and Wong (2002)<br>Govers and Krop (1998) | V<br>V<br>V<br>V<br>Q<br>Q<br>Q<br>Q | 186<br><br><br><br><br><br>535 |
| 1,2,3,6,7-pentachlorodibenzo-$p$-dioxin<br>$C_{12}H_3Cl_5O_2$<br>(PCDD-12367)<br>[71925-15-0]<br>RLGWDUHOIIWPGN-UHFFFAOYSA-N | 1.5<br><br>$7.8\times10^{-1}$ | | Wang and Wong (2002)<br><br>Govers and Krop (1998) | Q<br><br>Q | 535 |



Table A6.7: Polychlorinated dibenzo-$p$-dioxins (PCDDs) (. . . continued)

| Substance<br>Formula<br>(Trivial Name)<br>[CAS Registry Number]<br>InChIKey | $H_s^{cp}$<br>(at $T^{\ominus}$)<br>$\left[\dfrac{\text{mol}}{\text{m}^3\,\text{Pa}}\right]$ | $\dfrac{\text{d}\ln H_s^{cp}}{\text{d}(1/T)}$<br><br>[K] | Reference | Type | Note |
|---|---|---|---|---|---|
| 1,2,3,6,8-pentachlorodibenzo-$p$-dioxin<br>$C_{12}H_3Cl_5O_2$<br>(PCDD-12368)<br>[71925-16-1]<br>VKDGHBBUEIIEHL-UHFFFAOYSA-N | 1.3<br>$9.5\times10^{-1}$ | | Wang and Wong (2002)<br>Govers and Krop (1998) | Q<br>Q | 535 |
| 1,2,3,6,9-pentachlorodibenzo-$p$-dioxin<br>$C_{12}H_3Cl_5O_2$<br>(PCDD-12369)<br>[82291-34-7]<br>NWKWRHSKKNELND-UHFFFAOYSA-N | 1.4<br>1.9 | | Wang and Wong (2002)<br>Govers and Krop (1998) | Q<br>Q | 535 |
| 1,2,3,7,8-pentachlorodibenzo-$p$-dioxin<br>$C_{12}H_3Cl_5O_2$<br>(PCDD-12378)<br>[40321-76-4]<br>FSPZPQQWDODWAU-UHFFFAOYSA-N | $5.2\times10^{-5}$<br>$6.4\times10^{-1}$<br>1.5<br>$6.8\times10^{-1}$ | 2500 | Paasivirta et al. (1999)<br>Saçan et al. (2005)<br>Wang and Wong (2002)<br>Govers and Krop (1998) | T<br>Q<br>Q<br>Q | <br><br>535 |
| 1,2,3,7,9-pentachlorodibenzo-$p$-dioxin<br>$C_{12}H_3Cl_5O_2$<br>(PCDD-12379)<br>[71925-17-2]<br>UAOYHTXYVWEPIB-UHFFFAOYSA-N | 1.3<br>1.5 | | Wang and Wong (2002)<br>Govers and Krop (1998) | Q<br>Q | 535 |
| 1,2,3,8,9-pentachlorodibenzo-$p$-dioxin<br>$C_{12}H_3Cl_5O_2$<br>(PCDD-12389)<br>[71925-18-3]<br>VUMZAVNIADYKFC-UHFFFAOYSA-N | 1.5<br>1.4 | | Wang and Wong (2002)<br>Govers and Krop (1998) | Q<br>Q | 535 |
| 1,2,4,6,7-pentachlorodibenzo-$p$-dioxin<br>$C_{12}H_3Cl_5O_2$<br>(PCDD-12467)<br>[82291-35-8]<br>SEKDDGLKEYEVQK-UHFFFAOYSA-N | 1.4<br>1.4 | | Wang and Wong (2002)<br>Govers and Krop (1998) | Q<br>Q | 535 |
| 1,2,4,6,8-pentachlorodibenzo-$p$-dioxin<br>$C_{12}H_3Cl_5O_2$<br>(PCDD-12468)<br>[71998-76-0]<br>SJJWALZHAWITMS-UHFFFAOYSA-N | 1.2<br>2.1 | | Wang and Wong (2002)<br>Govers and Krop (1998) | Q<br>Q | 535 |



Table A6.7: Polychlorinated dibenzo-$p$-dioxins (PCDDs) (...continued)

| Substance Formula (Trivial Name) [CAS Registry Number] InChIKey | $H_s^{cp}$ (at $T^\ominus$) $\left[\dfrac{\text{mol}}{\text{m}^3\,\text{Pa}}\right]$ | $\dfrac{\text{d}\ln H_s^{cp}}{\text{d}(1/T)}$ [K] | Reference | Type | Note |
|---|---|---|---|---|---|
| 1,2,4,6,9-pentachlorodibenzo-$p$-dioxin | 1.3 | | Wang and Wong (2002) | Q | 535 |
| $C_{12}H_3Cl_5O_2$ | 3.6 | | Govers and Krop (1998) | Q | |
| (PCDD-12469) | | | | | |
| [82291-36-9] | | | | | |
| GNQVSAMSAKZLKE-UHFFFAOYSA-N | | | | | |
| 1,2,4,7,8-pentachlorodibenzo-$p$-dioxin | 1.3 | | Wang and Wong (2002) | Q | 535 |
| $C_{12}H_3Cl_5O_2$ | $9.1\times10^{-1}$ | | Govers and Krop (1998) | Q | |
| (PCDD-12478) | | | | | |
| [58802-08-7] | | | | | |
| QUPLGUUISJOUPJ-UHFFFAOYSA-N | | | | | |
| 1,2,4,7,9-pentachlorodibenzo-$p$-dioxin | 1.2 | | Wang and Wong (2002) | Q | 535 |
| $C_{12}H_3Cl_5O_2$ | 1.8 | | Govers and Krop (1998) | Q | |
| (PCDD-12479) | | | | | |
| [82291-37-0] | | | | | |
| QLBBXWPVEFJZEC-UHFFFAOYSA-N | | | | | |
| 1,2,4,8,9-pentachlorodibenzo-$p$-dioxin | 1.4 | | Wang and Wong (2002) | Q | 535 |
| $C_{12}H_3Cl_5O_2$ | 1.9 | | Govers and Krop (1998) | Q | |
| (PCDD-12489) | | | | | |
| [82291-38-1] | | | | | |
| KLLFLRKEOJCTGC-UHFFFAOYSA-N | | | | | |
| 1,2,3,4,6,7-hexachlorodibenzo-$p$-dioxin | 2.5 | | Wang and Wong (2002) | Q | 535 |
| $C_{12}H_2Cl_6O_2$ | 1.5 | | Govers and Krop (1998) | Q | |
| (PCDD-123467) | | | | | |
| [58200-66-1] | | | | | |
| NLBQVWJHLWAFGJ-UHFFFAOYSA-N | | | | | |
| 1,2,3,4,6,8-hexachlorodibenzo-$p$-dioxin | 2.2 | | Wang and Wong (2002) | Q | 535 |
| $C_{12}H_2Cl_6O_2$ | 1.8 | | Govers and Krop (1998) | Q | |
| (PCDD-123468) | | | | | |
| [58200-67-2] | | | | | |
| IMALTUQZEIFHJW-UHFFFAOYSA-N | | | | | |
| 1,2,3,4,6,9-hexachlorodibenzo-$p$-dioxin | 2.3 | | Wang and Wong (2002) | Q | 535 |
| $C_{12}H_2Cl_6O_2$ | 4.0 | | Govers and Krop (1998) | Q | |
| (PCDD-123469) | | | | | |
| [58200-68-3] | | | | | |
| UDYXCMRDCOVQLG-UHFFFAOYSA-N | | | | | |



Table A6.7: Polychlorinated dibenzo-$p$-dioxins (PCDDs) (...continued)

| Substance Formula (Trivial Name) [CAS Registry Number] InChIKey | $H_s^{cp}$ (at $T^{\ominus}$) $\left[\dfrac{\mathrm{mol}}{\mathrm{m^3\,Pa}}\right]$ | $\dfrac{\mathrm{d}\ln H_s^{cp}}{\mathrm{d}(1/T)}$ [K] | Reference | Type | Note |
|---|---|---|---|---|---|
| 1,2,3,4,7,8-hexachlorodibenzo-$p$-dioxin | 2.5 | | Duchowicz et al. (2020) | V | 186 |
| $C_{12}H_2Cl_6O_2$ | 3.0 | | Mackay et al. (2006b) | V | |
| (PCDD-123478) | 1.6 | | Govers and Krop (1998) | V | |
| [39227-28-6] | $2.2\times10^{-1}$ | | Shiu et al. (1988) | V | |
| WCYYQNSQJHPVMG-UHFFFAOYSA-N | $1.2\times10^{-4}$ | 2900 | Paasivirta et al. (1999) | T | |
| | $3.7\times10^{-1}$ | | Duchowicz et al. (2020) | Q | |
| | | 8800 | Kühne et al. (2005) | Q | |
| | $7.7\times10^{-1}$ | | Saçan et al. (2005) | Q | |
| | 2.3 | | Wang and Wong (2002) | Q | 535 |
| | $6.9\times10^{-1}$ | | Govers and Krop (1998) | Q | |
| | | 9400 | Kühne et al. (2005) | ? | |
| 1,2,3,6,7,8-hexachlorodibenzo-$p$-dioxin | $6.2\times10^{-5}$ | 2800 | Paasivirta et al. (1999) | T | |
| $C_{12}H_2Cl_6O_2$ | 5.2 | | HSDB (2015) | Q | 99 |
| (PCDD-123678) | $7.4\times10^{-1}$ | | Saçan et al. (2005) | Q | |
| [57653-85-7] | 2.4 | | Wang and Wong (2002) | Q | 535 |
| YCLUIPQDHHPDJJ-UHFFFAOYSA-N | $6.9\times10^{-1}$ | | Govers and Krop (1998) | Q | |
| 1,2,3,6,7,9-hexachlorodibenzo-$p$-dioxin | 2.2 | | Wang and Wong (2002) | Q | 535 |
| $C_{12}H_2Cl_6O_2$ | 1.7 | | Govers and Krop (1998) | Q | |
| (PCDD-123679) | | | | | |
| [64461-98-9] | | | | | |
| BQOHWGKNRKCEFT-UHFFFAOYSA-N | | | | | |
| 1,2,3,6,8,9-hexachlorodibenzo-$p$-dioxin | 2.2 | | Wang and Wong (2002) | Q | 535 |
| $C_{12}H_2Cl_6O_2$ | 1.8 | | Govers and Krop (1998) | Q | |
| (PCDD-123689) | | | | | |
| [58200-69-4] | | | | | |
| GZRQZUFXVFRKBI-UHFFFAOYSA-N | | | | | |
| 1,2,3,7,8,9-hexachlorodibenzo-$p$-dioxin | $2.5\times10^{-4}$ | 2700 | Paasivirta et al. (1999) | T | |
| $C_{12}H_2Cl_6O_2$ | 5.2 | | HSDB (2015) | Q | 99 |
| (PCDD-123789) | 1.1 | | Saçan et al. (2005) | Q | |
| [19408-74-3] | 2.4 | | Wang and Wong (2002) | Q | 535 |
| LGIRBUBHIWTVCK-UHFFFAOYSA-N | 1.2 | | Govers and Krop (1998) | Q | |
| 1,2,4,6,7,9-hexachlorodibenzo-$p$-dioxin | 2.1 | | Wang and Wong (2002) | Q | 535 |
| $C_{12}H_2Cl_6O_2$ | 3.5 | | Govers and Krop (1998) | Q | |
| (PCDD-124679) | | | | | |
| [39227-62-8] | | | | | |
| BSJDQMWAWFTDGD-UHFFFAOYSA-N | | | | | |





Table A6.7: Polychlorinated dibenzo-$p$-dioxins (PCDDs) (... continued)

| Substance<br>Formula<br>(Trivial Name)<br>[CAS Registry Number]<br>InChIKey | $H_s^{cp}$<br>(at $T^{\ominus}$)<br>$\left[\dfrac{\text{mol}}{\text{m}^3\,\text{Pa}}\right]$ | $\dfrac{\mathrm{d}\ln H_s^{cp}}{\mathrm{d}(1/T)}$<br><br>[K] | Reference | Type | Note |
|---|---|---|---|---|---|
| 1,2,4,6,8,9-hexachlorodibenzo-$p$-dioxin | 2.0 | | Wang and Wong (2002) | Q | 535 |
| $C_{12}H_2Cl_6O_2$ | 4.4 | | Govers and Krop (1998) | Q | |
| (PCDD-124689) | | | | | |
| [58802-09-8] | | | | | |
| URELDHWUZUWPIU-UHFFFAOYSA-N | | | | | |
| 1,2,3,4,6,7,8-heptachlorodibenzo-$p$-dioxin | $5.6\times10^{-2}$ | | Duchowicz et al. (2020) | V | 186 |
| $C_{12}HCl_7O_2$ | 7.5 | | Mackay et al. (2006b) | V | |
| (PCDD-1234678) | 2.3 | | Govers and Krop (1998) | V | |
| [35822-46-9] | 7.5 | | Shiu et al. (1988) | V | |
| WCLNVRQZUKYVAI-UHFFFAOYSA-N | $7.5\times10^{-5}$ | 2400 | Paasivirta et al. (1999) | T | |
| | $4.5\times10^{-1}$ | | Duchowicz et al. (2020) | Q | |
| | $4.5\times10^{-1}$ | | HSDB (2015) | Q | 545 |
| | 1.4 | | Saçan et al. (2005) | Q | |
| | 3.6 | | Wang and Wong (2002) | Q | 535 |
| | 1.2 | | Govers and Krop (1998) | Q | |
| 1,2,3,4,6,7,9-heptachlorodibenzo-$p$-dioxin | 3.4 | | Wang and Wong (2002) | Q | 535 |
| $C_{12}HCl_7O_2$ | 3.2 | | Govers and Krop (1998) | Q | |
| (PCDD-1234679) | | | | | |
| [58200-70-7] | | | | | |
| KTJJIBIRZGQFQZ-UHFFFAOYSA-N | | | | | |
| octachlorodibenzo-$p$-dioxin | 1.5 | | Duchowicz et al. (2020) | V | 186 |
| $C_{12}Cl_8O_2$ | 1.5 | | HSDB (2015) | V | |
| (PCDD-12346789) | 1.5 | | Mackay et al. (2006b) | V | |
| [3268-87-9] | $7.6\times10^{-1}$ | | Govers and Krop (1998) | V | |
| FOIBFBMSLDGNHL-UHFFFAOYSA-N | 1.5 | | Shiu et al. (1988) | V | |
| | $1.1\times10^{-5}$ | 2300 | Paasivirta et al. (1999) | T | |
| | $3.8\times10^{-1}$ | | Duchowicz et al. (2020) | Q | |
| | | 9600 | Kühne et al. (2005) | Q | |
| | 1.7 | | Saçan et al. (2005) | Q | |
| | 5.2 | | Wang and Wong (2002) | Q | 535 |
| | 1.9 | | Govers and Krop (1998) | Q | |
| | | 9500 | Kühne et al. (2005) | ? | |



### A6.8 Chlorocarbons with nitrogen (C, H, O, N, Cl)

Table A6.8: Chlorocarbons with nitrogen (C, H, O, N, Cl)

| Substance<br>Formula<br>(Trivial Name)<br>[CAS Registry Number]<br>InChIKey | $H_s^{cp}$<br>(at $T^{\ominus}$)<br>$\left[\dfrac{\text{mol}}{\text{m}^3\,\text{Pa}}\right]$ | $\dfrac{\text{d}\ln H_s^{cp}}{\text{d}(1/T)}$<br><br>[K] | Reference | Type | Note |
|---|---|---|---|---|---|
| cyanogen chloride<br>NCCl<br>[506-77-4]<br>QPJDMGCKMHUXFD-UHFFFAOYSA-N | $1.2\times10^{-2}$<br>$5.1\times10^{-3}$ | | Hilal et al. (2008)<br>Yaws (1999) | Q<br>? | <br>21 |
| N,N-dichloromethylamine<br>$CH_3NCl_2$<br>[7651-91-4]<br>DWEYPOWXNBAIDU-UHFFFAOYSA-N | $3.3\times10^{-3}$<br>$3.3\times10^{-3}$<br>$3.3\times10^{-3}$ | 4300<br>4300<br>4300 | Burkholder et al. (2019)<br>Burkholder et al. (2015)<br>Cimetiere and de Laat (2009) | L<br>L<br>M | |
| chloroacetonitrile<br>$C_2H_2ClN$<br>[107-14-2]<br>RENMDAKOXSCIGH-UHFFFAOYSA-N | $9.1\times10^{-1}$ | <br>4600<br>5400 | HSDB (2015)<br>Kühne et al. (2005)<br>Kühne et al. (2005) | Q<br>Q<br>? | 447 |
| dichloroacetonitrile<br>$C_2HCl_2N$<br>[3018-12-0]<br>STZZWJCGRKXEFF-UHFFFAOYSA-N | 2.6 | | HSDB (2015) | Q | 99 |
| trichloroacetonitrile<br>$C_2Cl_3N$<br>[545-06-2]<br>DRUIESSIVFYOMK-UHFFFAOYSA-N | 7.6<br>7.3<br>$1.9\times10^{-2}$<br>$3.9\times10^{-3}$<br>$1.0\times10^{-2}$ | | HSDB (2015)<br>Zhang et al. (2010)<br>Zhang et al. (2010)<br>Zhang et al. (2010)<br>Zhang et al. (2010) | Q<br>Q<br>Q<br>Q<br>Q | 99<br>287, 288<br>287, 289<br>287, 290<br>287, 291 |
| tetramethylammonium chloride<br>$C_4H_{12}ClN$<br>[75-57-0]<br>OKIZCWYLBDKLSU-UHFFFAOYSA-M | $2.3\times10^{6}$ | | HSDB (2015) | Q | 99 |
| metformin hydrochloride<br>$C_4H_{12}ClN_5$<br>[1115-70-4]<br>OETHQSJEHLVLGH-UHFFFAOYSA-N | $1.3\times10^{10}$ | | HSDB (2015) | Q | 99 |
| bis(2-chloroethyl)methylamine<br>$C_5H_{11}Cl_2N$<br>[51-75-2]<br>HAWPXGHAZFHHAD-UHFFFAOYSA-N | 3.4<br>$8.5\times10^{-1}$ | | Duchowicz et al. (2020)<br>Duchowicz et al. (2020) | V<br>Q | 186 |
| chlormequat chloride<br>$C_5H_{13}Cl_2N$<br>[999-81-5]<br>OLQFELZRGYJRAZ-UHFFFAOYSA-N | $>3.2\times10^{9}$ | | Maniere et al. (2011) | ? | 241, 165 |



Table A6.8: Chlorocarbons with nitrogen (C, H, O, N, Cl) (... continued)

| Substance Formula (Trivial Name) [CAS Registry Number] InChIKey | $H_s^{cp}$ (at $T^{\ominus}$) $\left[\dfrac{\text{mol}}{\text{m}^3\,\text{Pa}}\right]$ | $\dfrac{\text{d}\ln H_s^{cp}}{\text{d}(1/T)}$ [K] | Reference | Type | Note |
|---|---|---|---|---|---|
| tris(2-chloroethyl)amine $C_6H_{12}Cl_3N$ [555-77-1] FDAYLTPAFBGXAB-UHFFFAOYSA-N | $5.3\times10^{-1}$ $5.3\times10^{-1}$ $7.3\times10^{-1}$ | | Duchowicz et al. (2020) HSDB (2015) Duchowicz et al. (2020) | V V Q | 186 |
| bis(2-chloroethyl)ethylamine $C_6H_{13}Cl_2N$ (ethylbis(2-chloroethyl)amine) [538-07-8] UQZPGHOJMQTOHB-UHFFFAOYSA-N | $2.8\times10^{-2}$ $2.9\times10^{-2}$ $3.0\times10^{-1}$ | | Duchowicz et al. (2020) HSDB (2015) Duchowicz et al. (2020) | V V Q | 186 |
| cetrimonium chloride $C_{19}H_{42}ClN$ (trimethylhexadecylammonium chloride) [112-02-7] WOWHHFRSBJGXCM-UHFFFAOYSA-M | $3.4\times10^{4}$ | | HSDB (2015) | Q | 99 |
| dimethyldioctadecylammonium chloride $C_{38}H_{80}ClN$ [107-64-2] REZZEXDLIUJMMS-UHFFFAOYSA-M | $1.5\times10^{2}$ | | HSDB (2015) | Q | 99 |
| 1-amino-2-chlorobenzene $C_6H_6ClN$ (o-chloroaniline) [95-51-2] AKCRQHGQIJBRMN-UHFFFAOYSA-N | 1.9 $7.8\times10^{-1}$ 1.8 $7.6\times10^{-1}$ 1.8 1.3 1.3 1.3 2.4 1.6 7.4 2.3 $9.6\times10^{1}$ 2.8 $1.1\times10^{1}$ 5.4 7.0 1.1 | 7900 | Brockbank (2013) Chao et al. (2017) Duchowicz et al. (2020) Chao et al. (2017) HSDB (2015) Mackay et al. (2006d) Lide and Frederikse (1995) Mackay et al. (1995) Meylan and Howard (1991) Abraham et al. (1994a) Duchowicz et al. (2020) Hilal et al. (2008) Modarresi et al. (2007) English and Carroll (2001) Katritzky et al. (1998) Nirmalakhandan et al. (1997) Meylan and Howard (1991) Yaws (1999) | L M V V V V V V V R Q Q Q Q Q Q Q ? | 1, 735 186 67 230, 231 21, 12 |
| 1-amino-3-chlorobenzene $C_6H_6ClN$ (m-chloroaniline) [108-42-9] PNPCRKVUWYDDST-UHFFFAOYSA-N | 4.7 9.8 2.7 4.5 4.5 7.5 7.5 | | Chao et al. (2017) Altschuh et al. (1999) Chao et al. (2017) Mackay et al. (2006d) Mackay et al. (1995) Abraham et al. (1994a) Yaws (2003) | M M V V V R X | 237, 12 |



Table A6.8: Chlorocarbons with nitrogen (C, H, O, N, Cl) (. . . continued)

| Substance Formula (Trivial Name) [CAS Registry Number] InChIKey | $H_s^{cp}$ (at $T^\ominus$) $\left[\dfrac{\mathrm{mol}}{\mathrm{m^3\,Pa}}\right]$ | $\dfrac{\mathrm{d}\ln H_s^{cp}}{\mathrm{d}(1/T)}$ [K] | Reference | Type | Note |
|---|---|---|---|---|---|
| | 9.1 | | Keshavarz et al. (2022) | Q | |
| | 8.8 | | Duchowicz et al. (2020) | Q | |
| | 4.9 | | Gharagheizi et al. (2010) | Q | 246 |
| | 7.7 | | Hilal et al. (2008) | Q | |
| | $3.1\times10^1$ | | Modarresi et al. (2007) | Q | 67 |
| | 4.6 | | Yao et al. (2002) | Q | 229 |
| | 4.1 | | English and Carroll (2001) | Q | 230, 274 |
| | $1.5\times10^1$ | | Katritzky et al. (1998) | Q | |
| | 5.3 | | Nirmalakhandan et al. (1997) | Q | |
| | 9.9 | | Duchowicz et al. (2020) | ? | 185, 21 |
| | 4.9 | | Yaws (1999) | ? | 21, 12 |
| 1-amino-4-chlorobenzene C$_6$H$_6$ClN (*p*-chloroaniline) [106-47-8] QSNSCYSYFYORTR-UHFFFAOYSA-N | 8.5 | | Duchowicz et al. (2020) | V | 186 |
| | 3.2 | | HSDB (2015) | V | |
| | $1.0\times10^1$ | | Mackay et al. (2006d) | V | |
| | $9.1\times10^{-1}$ | | Lide and Frederikse (1995) | V | |
| | $1.0\times10^1$ | | Mackay et al. (1995) | V | |
| | $2.5\times10^1$ | | Meylan and Howard (1991) | V | |
| | 8.6 | | Abraham et al. (1994a) | R | |
| | 3.4 | | Yaws (2003) | X | 237, 12 |
| | $9.2\times10^{-1}$ | | Howard (1989) | X | 412 |
| | 8.3 | | Duchowicz et al. (2020) | Q | |
| | 4.9 | | Gharagheizi et al. (2010) | Q | 246 |
| | 8.6 | | Hilal et al. (2008) | Q | |
| | $1.6\times10^1$ | | Modarresi et al. (2007) | Q | 67 |
| | 8.8 | | Yaffe et al. (2003) | Q | 248, 249 |
| | 5.4 | | English and Carroll (2001) | Q | 230, 260 |
| | $1.5\times10^1$ | | Katritzky et al. (1998) | Q | |
| | 5.3 | | Nirmalakhandan et al. (1997) | Q | |
| | 7.0 | | Meylan and Howard (1991) | Q | |
| 2,3-dichlorobenzenamine C$_6$H$_5$Cl$_2$N (2,3-dichloroaniline) [608-27-5] BRPSAOUFIJSKOT-UHFFFAOYSA-N | 6.2 | | HSDB (2015) | Q | 99 |
| 2,4-dichlorobenzenamine C$_6$H$_5$Cl$_2$N (2,4-dichloroaniline) [554-00-7] KQCMTOWTPBNWDB-UHFFFAOYSA-N | 6.2 | | HSDB (2015) | Q | 99 |
| 3,4-dichlorobenzenamine C$_6$H$_5$Cl$_2$N (3,4-dichloroaniline) [95-76-1] SDYWXFYBZPNOFX-UHFFFAOYSA-N | $6.8\times10^{-1}$ | | Duchowicz et al. (2020) | V | 186 |
| | $6.8\times10^{-1}$ | | HSDB (2015) | V | |
| | $4.4\times10^{-1}$ | | Mackay et al. (2006d) | V | |
| | $4.4\times10^{-1}$ | | Mackay et al. (1995) | V | |
| | $1.8\times10^1$ | | Duchowicz et al. (2020) | Q | |
| | $2.0\times10^1$ | | Katritzky et al. (1998) | Q | |



Table A6.8: Chlorocarbons with nitrogen (C, H, O, N, Cl) (... continued)

| Substance Formula (Trivial Name) [CAS Registry Number] InChIKey | $H_s^{cp}$ (at $T^\ominus$) $\left[\dfrac{\mathrm{mol}}{\mathrm{m^3\,Pa}}\right]$ | $\dfrac{\mathrm{d}\ln H_s^{cp}}{\mathrm{d}(1/T)}$ [K] | Reference | Type | Note |
|---|---|---|---|---|---|
| 3,5-dichlorobenzenamine $C_6H_5Cl_2N$ (3,5-dichloroaniline) [626-43-7] UQRLKWGPEVNVHT-UHFFFAOYSA-N | 6.2 | | HSDB (2015) | Q | 99 |
| 2,5-dichlorobenzenamine $C_6H_5Cl_2N$ (2,5-dichloroaniline) [95-82-9] AVYGCQXNNJPXSS-UHFFFAOYSA-N | 6.2 9.5 2.6 2.7 $1.9\times10^1$ | | HSDB (2015) Zhang et al. (2010) Zhang et al. (2010) Zhang et al. (2010) Zhang et al. (2010) | Q Q Q Q Q | 99 287, 288 287, 289 287, 290 287, 291 |
| 2,4,5-trichlorobenzenamine $C_6H_4Cl_3N$ [636-30-6] GUMCAKKKNKYFEB-UHFFFAOYSA-N | $1.3\times10^1$ 2.4 9.5 $1.8\times10^1$ | | Zhang et al. (2010) Zhang et al. (2010) Zhang et al. (2010) Zhang et al. (2010) | Q Q Q Q | 287, 288 287, 289 287, 290 287, 291 |
| 2,4,6-trichlorobenzenamine $C_6H_4Cl_3N$ [634-93-5] NATVSFWWYVJTAZ-UHFFFAOYSA-N | 7.4 $1.3\times10^1$ $6.2\times10^{-1}$ $4.1\times10^{-1}$ $1.5\times10^1$ | | HSDB (2015) Zhang et al. (2010) Zhang et al. (2010) Zhang et al. (2010) Zhang et al. (2010) | V Q Q Q Q | 287, 288 287, 289 287, 290 287, 291 |
| 2,3,4,5,6-pentachloroaniline $C_6H_2Cl_5N$ [527-20-8] KHCZSJXTDDHLGJ-UHFFFAOYSA-N | $2.3\times10^1$ | | HSDB (2015) | Q | 99 |
| 2,6-dichlorobenzenenitrile $C_6H_3Cl_2CN$ (dichlobenil) [1194-65-6] YOYAIZYFCNQIRF-UHFFFAOYSA-N | $4.8\times10^{-1}$ $9.8\times10^{-1}$ $9.9\times10^{-1}$ 1.5 1.4 1.5 1.4 $1.5\times10^{-2}$ $3.5\times10^{-1}$ $1.5\times10^{-2}$ $3.6\times10^{-1}$ | 5400 6000 5500 | Schoene and Steinhanses (1985) Duchowicz et al. (2020) HSDB (2015) Mackay et al. (2006d) Schüürmann (2000) Suntio et al. (1988) Burkhard and Guth (1981) Barcelo and Hennion (1997) Duchowicz et al. (2020) Goodarzi et al. (2010) Modarresi et al. (2007) Kühne et al. (2005) Kühne et al. (2005) | M V V V V V V X Q Q Q Q ? | 186 12 567 568, 569 67 |
| (2,4,6-trichlorophenyl)hydrazine $C_6H_5Cl_3N_2$ [5329-12-4] MULHANRBCQBHII-UHFFFAOYSA-N | $3.1\times10^3$ $3.7\times10^1$ $1.1\times10^1$ $5.4\times10^3$ | | Zhang et al. (2010) Zhang et al. (2010) Zhang et al. (2010) Zhang et al. (2010) | Q Q Q Q | 287, 288 287, 289 287, 290 287, 291 |



Table A6.8: Chlorocarbons with nitrogen (C, H, O, N, Cl) (...continued)

| Substance Formula (Trivial Name) [CAS Registry Number] InChIKey | $H_s^{cp}$ (at $T^{\ominus}$) $\left[\dfrac{\mathrm{mol}}{\mathrm{m^3\,Pa}}\right]$ | $\dfrac{\mathrm{d}\ln H_s^{cp}}{\mathrm{d}(1/T)}$ [K] | Reference | Type | Note |
|---|---|---|---|---|---|
| 4-chlorobenzonitrile C$_7$H$_4$ClN [623-03-0] GJNGXPDXRVXSEH-UHFFFAOYSA-N | $2.5\times10^{-1}$ $3.8\times10^{-1}$ 1.6 $4.5\times10^{-1}$ | | Zhang et al. (2010) Zhang et al. (2010) Zhang et al. (2010) Zhang et al. (2010) | Q Q Q Q | 287, 288 287, 289 287, 290 287, 291 |
| 3-chloro-2-methylbenzenamine C$_7$H$_8$ClN [87-60-5] ZUVPLKVDZNDZCM-UHFFFAOYSA-N | 6.3 | | HSDB (2015) | Q | 99 |
| 3-chloro-4-methylbenzenamine C$_7$H$_8$ClN [95-74-9] RQKFYFNZSHWXAW-UHFFFAOYSA-N | 4.9 | | HSDB (2015) | Q | 99 |
| 4-chloro-2-methylbenzenamine C$_7$H$_8$ClN [95-69-2] CXNVOWPRHWWCQR-UHFFFAOYSA-N | 4.9 | | HSDB (2015) | Q | 99 |
| 5-chloro-2-methylbenzenamine C$_7$H$_8$ClN [95-79-4] WRZOMWDJOLIVQP-UHFFFAOYSA-N | 6.3 | | HSDB (2015) | Q | 545 |
| 2,4,5,6-tetrachloro-1,3-dicyanobenzene C$_8$Cl$_4$N$_2$ (chlorothalonil) [1897-45-6] CRQQGFGUEAVUIL-UHFFFAOYSA-N | $5.0\times10^{1}$ $1.7\times10^{-2}$ $3.9\times10^{1}$ $4.5\times10^{1}$ $4.1\times10^{-1}$ $1.8\times10^{1}$ $6.5\times10^{1}$ 1.5 $6.9\times10^{1}$ $2.7\times10^{1}$ 5.8 $2.4\times10^{1}$ $6.5\times10^{1}$ 4.9 | | Kawamoto and Urano (1989) Mackay et al. (2006d) MacBean (2012b) Armbrust (2000) Keshavarz et al. (2022) Duchowicz et al. (2020) Zhang et al. (2010) Zhang et al. (2010) Zhang et al. (2010) Zhang et al. (2010) Hilal et al. (2008) Modarresi et al. (2007) Meylan and Howard (1991) Duchowicz et al. (2020) | M V X C Q Q Q Q Q Q Q Q Q ? | 350 287, 288 287, 289 287, 290 287, 291 67 185, 21 |
| 2-chlorobenzalmalononitrile C$_{10}$H$_5$ClN$_2$ [2698-41-1] JJNZXLAFIPKXIG-UHFFFAOYSA-N | $9.9\times10^{2}$ | | HSDB (2015) | Q | 99 |





Table A6.8: Chlorocarbons with nitrogen (C, H, O, N, Cl) (...continued)

| Substance Formula (Trivial Name) [CAS Registry Number] InChIKey | $H_s^{cp}$ (at $T^\ominus$) $\left[\dfrac{\text{mol}}{\text{m}^3\,\text{Pa}}\right]$ | $\dfrac{\text{d}\ln H_s^{cp}}{\text{d}(1/T)}$ [K] | Reference | Type | Note |
|---|---|---|---|---|---|
| benzyltrimethylammonium chloride $C_{10}H_{16}ClN$ [56-93-9] KXHPPCXNWTUNSB-UHFFFAOYSA-M | $2.9\times10^8$ | | HSDB (2015) | Q | 99 |
| chlordimeform $C_{10}H_{13}ClN_2$ [6164-98-3] STUSTWKEFDQFFZ-UHFFFAOYSA-N | $2.9\times10^1$ $2.6\times10^1$ | | HSDB (2015) MacBean (2012a) | V ? | 12 |
| 4,4'-dichloroazobenzene $C_{12}H_8Cl_2N_2$ [1602-00-2] XHQLXCFUPJSGOE-UHFFFAOYSA-N | 1.2 | | HSDB (2015) | Q | 99 |
| bis(3,4-dichlorophenyl)diazene $C_{12}H_6Cl_4N_2$ (3,4,3',4'-tetrachloroazobenzene) [14047-09-7] SOBGIMQKWDUEPY-UHFFFAOYSA-N | 2.2 | | HSDB (2015) | Q | 447 |
| 2-(*p*-chlorophenyl)-3-methylbutyronitrile $C_{11}H_{12}ClN$ [2012-81-9] RBGSZIRWNWQDOK-UHFFFAOYSA-N | 2.3 4.4 3.9 $2.9\times10^{-1}$ | | Zhang et al. (2010) Zhang et al. (2010) Zhang et al. (2010) Zhang et al. (2010) | Q Q Q Q | 287, 288 287, 289 287, 290 287, 291 |
| 3,3'-dichloro-(1,1'-biphenyl)-4,4'-diamine $C_{12}H_{10}Cl_2N_2$ (3,3'-dichlorobenzidine) [91-94-1] HUWXDEQWWKGHRV-UHFFFAOYSA-N | $2.0\times10^2$ $2.0\times10^2$ $1.2\times10^1$ $3.5\times10^5$ | | Mackay et al. (2006d) Mackay et al. (1995) Mackay et al. (1995) HSDB (2015) | V V C Q | 99 |
| 4-chloroazobenzene $C_{12}H_9ClN_2$ [4340-77-6] NJFDMENHTAYHMA-UHFFFAOYSA-N | 7.4 2.2 | | Duchowicz et al. (2020) Duchowicz et al. (2020) | V Q | 186 |
| 4,4'-methylenebis(2-chlorobenzenamine) $C_{13}H_{12}Cl_2N_2$ [101-14-4] IBOFVQJTBBUKMU-UHFFFAOYSA-N | $9.0\times10^5$ $3.0\times10^5$ $3.4\times10^4$ $2.9\times10^4$ $9.7\times10^5$ | | HSDB (2015) Zhang et al. (2010) Zhang et al. (2010) Zhang et al. (2010) Zhang et al. (2010) | Q Q Q Q Q | 99 287, 288 287, 289 287, 290 287, 291 |
| clofentezine $C_{14}H_8N_4Cl_2$ [74115-24-5] UXADOQPNKNTIHB-UHFFFAOYSA-N | $2.5\times10^4$ $1.7\times10^4$ | | Duchowicz et al. (2020) Duchowicz et al. (2020) | V Q | 186 |





Table A6.8: Chlorocarbons with nitrogen (C, H, O, N, Cl) (...continued)

| Substance Formula (Trivial Name) [CAS Registry Number] InChIKey | $H_s^{cp}$ (at $T^\ominus$) $\left[\dfrac{\text{mol}}{\text{m}^3\,\text{Pa}}\right]$ | $\dfrac{\text{d}\ln H_s^{cp}}{\text{d}(1/T)}$ [K] | Reference | Type | Note |
|---|---|---|---|---|---|
| aniline, 4,4'-(imidocarbonyl)bis-(N,N-dimethyl)-, hydrochloride $C_{17}H_{22}ClN_3$ (auramine hydrochloride) [2465-27-2] KSCQDDRPFHTIRL-UHFFFAOYSA-N | $3.5\times10^{10}$ | | HSDB (2015) | Q | 99 |
| amitriptyline hydrochloride $C_{20}H_{24}ClN$ [549-18-8] KFYRPLNVJVHZGT-UHFFFAOYSA-N | $1.4\times10^{2}$ | | HSDB (2015) | Q | 99 |
| chlorhexidine $C_{22}H_{30}Cl_2N_{10}$ [55-56-1] GHXZTYHSJHQHIJ-UHFFFAOYSA-N | $9.0\times10^{24}$ | | HSDB (2015) | Q | 99 |
| malachite green $C_{23}H_{25}ClN_2$ [569-64-2] FDZZZRQASAIRJF-UHFFFAOYSA-M | $5.2\times10^{8}$ | | HSDB (2015) | Q | 99 |
| tetradecylbenzyldimethyl ammonium chloride $C_{23}H_{42}ClN$ [139-08-2] OCBHHZMJRVXXQK-UHFFFAOYSA-M | $7.6\times10^{5}$ | | HSDB (2015) | Q | 99 |
| stearyldimethylbenzylammonium chloride $C_{27}H_{50}ClN$ (benzyldimethylstearylammonium chloride) [122-19-0] SFVFIFLLYFPGHH-UHFFFAOYSA-M | $2.3\times10^{5}$ | | HSDB (2015) | Q | 99 |
| 2,4,6-trichloro-1,3,5-triazine $C_3Cl_3N_3$ [108-77-0] MGNCLNQXLYJVJD-UHFFFAOYSA-N | $2.0\times10^{1}$ $2.0\times10^{1}$ $2.4\times10^{1}$ $3.1\times10^{-1}$ $7.9$ | | HSDB (2015) Zhang et al. (2010) Zhang et al. (2010) Zhang et al. (2010) Zhang et al. (2010) | Q Q Q Q Q | 99 287, 288 287, 289 287, 290 287, 291 |
| 2-chloropyridine $C_5H_4ClN$ [109-09-1] OKDGRDCXVWSXDC-UHFFFAOYSA-N | $7.4\times10^{-1}$ $6.1\times10^{-1}$ $1.9\times10^{-1}$ $5.8\times10^{-1}$ $1.1$ $6.7\times10^{-1}$ | 5900 6100 | Arnett and Chawla (1979) Duchowicz et al. (2020) Duchowicz et al. (2020) Hilal et al. (2008) Modarresi et al. (2007) Kühne et al. (2005) Yaffe et al. (2003) | M V Q Q Q Q Q | 559 186 67 248, 249 |



Table A6.8: Chlorocarbons with nitrogen (C, H, O, N, Cl) (...continued)

| Substance Formula (Trivial Name) [CAS Registry Number] InChIKey | $H_s^{cp}$ (at $T^{\ominus}$) $\left[\dfrac{\mathrm{mol}}{\mathrm{m}^3\,\mathrm{Pa}}\right]$ | $\dfrac{\mathrm{d}\ln H_s^{cp}}{\mathrm{d}(1/T)}$ [K] | Reference | Type | Note |
|---|---|---|---|---|---|
| | $2.9\times10^{-1}$ | | English and Carroll (2001) | Q | 230, 274 |
| | $8.0\times10^{-2}$ | | Katritzky et al. (1998) | Q | |
| | $1.5\times10^{1}$ | | Nirmalakhandan et al. (1997) | Q | |
| | | 6600 | Kühne et al. (2005) | ? | |
| 3-chloropyridine $C_5H_4ClN$ [626-60-8] PWRBCZZQRRPXAB-UHFFFAOYSA-N | $3.5\times10^{-1}$ | 5600 | Arnett and Chawla (1979) | M | 559 |
| | $4.1\times10^{-1}$ | | Hilal et al. (2008) | Q | |
| | $8.3\times10^{-1}$ | | Modarresi et al. (2007) | Q | 67 |
| | $3.7\times10^{-1}$ | | Yaffe et al. (2003) | Q | 248, 249 |
| | $2.9\times10^{-1}$ | | English and Carroll (2001) | Q | 230, 231 |
| | $1.5\times10^{1}$ | | Nirmalakhandan et al. (1997) | Q | |
| 2,3,4,6-tetrachloropyridine $C_5HCl_4N$ [14121-36-9] FZFNDUTVMQOPCT-UHFFFAOYSA-N | $1.2\times10^{-3}$ | | Zhang et al. (2010) | Q | 287, 288 |
| | $1.1\times10^{-1}$ | | Zhang et al. (2010) | Q | 287, 289 |
| | $7.9\times10^{-2}$ | | Zhang et al. (2010) | Q | 287, 290 |
| | $2.9\times10^{-1}$ | | Zhang et al. (2010) | Q | 287, 291 |
| 2,3,5,6-tetrachloropyridine $C_5HCl_4N$ [2402-79-1] FATBKZJZAHWCSL-UHFFFAOYSA-N | $1.2\times10^{-3}$ | | HSDB (2015) | Q | 99 |
| | $1.2\times10^{-3}$ | | Zhang et al. (2010) | Q | 287, 288 |
| | $3.4\times10^{-2}$ | | Zhang et al. (2010) | Q | 287, 289 |
| | $8.4\times10^{-2}$ | | Zhang et al. (2010) | Q | 287, 290 |
| | $1.9\times10^{-1}$ | | Zhang et al. (2010) | Q | 287, 291 |
| pentachloropyridine $C_5Cl_5N$ [2176-62-7] DNDPLEAVNVOOQZ-UHFFFAOYSA-N | $1.6\times10^{-3}$ | | HSDB (2015) | Q | 99 |
| | $1.6\times10^{-3}$ | | Zhang et al. (2010) | Q | 287, 288 |
| | $1.3\times10^{-2}$ | | Zhang et al. (2010) | Q | 287, 289 |
| | $2.5\times10^{-2}$ | | Zhang et al. (2010) | Q | 287, 290 |
| | $1.8\times10^{-1}$ | | Zhang et al. (2010) | Q | 287, 291 |
| desethylatrazine $C_6H_{10}ClN_5$ [6190-65-4] DFWFIQKMSFGDCQ-UHFFFAOYSA-N | $6.6\times10^{3}$ | | HSDB (2015) | Q | 99 |
| 2-chloro-6-(trichloromethyl)-pyridine $C_6H_3Cl_4N$ [1929-82-4] DCUJJWWUNKIJPH-UHFFFAOYSA-N | $6.2\times10^{-1}$ | | Zhang et al. (2010) | Q | 287, 288 |
| | 1.8 | | Zhang et al. (2010) | Q | 287, 289 |
| | 2.7 | | Zhang et al. (2010) | Q | 287, 290 |
| | $4.2\times10^{-1}$ | | Zhang et al. (2010) | Q | 287, 291 |
| | 1.8 | | Hilal et al. (2008) | Q | |
| | $5.9\times10^{-1}$ | | Modarresi et al. (2007) | Q | 67 |
| 2,3,4,5-tetrachloro-6-methylpyridine $C_6H_3Cl_4N$ [10469-02-0] OZLPEWBQZDDFCQ-UHFFFAOYSA-N | $6.7\times10^{-2}$ | | Zhang et al. (2010) | Q | 287, 288 |
| | $3.6\times10^{-2}$ | | Zhang et al. (2010) | Q | 287, 289 |
| | $3.6\times10^{-2}$ | | Zhang et al. (2010) | Q | 287, 290 |
| | $1.1\times10^{-1}$ | | Zhang et al. (2010) | Q | 287, 291 |





Table A6.8: Chlorocarbons with nitrogen (C, H, O, N, Cl) (...continued)

| Substance<br>Formula<br>(Trivial Name)<br>[CAS Registry Number]<br>InChIKey | $H_s^{cp}$ (at $T^{\ominus}$) $\left[\dfrac{\mathrm{mol}}{\mathrm{m^3\,Pa}}\right]$ | $\dfrac{\mathrm{d}\ln H_s^{cp}}{\mathrm{d}(1/T)}$ [K] | Reference | Type | Note |
|---|---|---|---|---|---|
| 2-chloro-5-(trichloromethyl)pyridine | $6.2\times10^{-1}$ | | Zhang et al. (2010) | Q | 287, 288 |
| $C_6H_3Cl_4N$ | 2.2 | | Zhang et al. (2010) | Q | 287, 289 |
| [69045-78-9] | $7.7\times10^{-1}$ | | Zhang et al. (2010) | Q | 287, 290 |
| VLJIVLGVKMTBOD-UHFFFAOYSA-N | $4.2\times10^{-1}$ | | Zhang et al. (2010) | Q | 287, 291 |
| 2,3-dichloro-5-(trichloromethyl)pyridine | $8.4\times10^{-1}$ | | Zhang et al. (2010) | Q | 287, 288 |
| $C_6H_2Cl_5N$ | $4.1\times10^{-1}$ | | Zhang et al. (2010) | Q | 287, 289 |
| [69045-83-6] | $1.8\times10^{-1}$ | | Zhang et al. (2010) | Q | 287, 290 |
| XVBWGQSXLITICX-UHFFFAOYSA-N | $6.1\times10^{-1}$ | | Zhang et al. (2010) | Q | 287, 291 |
| 2,5-dichloro-6-(trichloromethyl)pyridine | $8.4\times10^{-1}$ | | Zhang et al. (2010) | Q | 287, 288 |
| $C_6H_2Cl_5N$ | 1.2 | | Zhang et al. (2010) | Q | 287, 289 |
| [1817-13-6] | $9.9\times10^{-1}$ | | Zhang et al. (2010) | Q | 287, 290 |
| MWFCRQNUHFSUNY-UHFFFAOYSA-N | $5.7\times10^{-1}$ | | Zhang et al. (2010) | Q | 287, 291 |
| 3,4,5-trichloro-2-(trichloromethyl)pyridine | $7.2\times10^{1}$ | | Zhang et al. (2010) | Q | 287, 288 |
| $C_6HCl_6N$ | $9.0\times10^{-2}$ | | Zhang et al. (2010) | Q | 287, 289 |
| [1201-30-5] | $1.0\times10^{-1}$ | | Zhang et al. (2010) | Q | 287, 290 |
| YWSFDYUMEBEQNZ-UHFFFAOYSA-N | $4.1\times10^{-1}$ | | Zhang et al. (2010) | Q | 287, 291 |
| 2,3,4,5-tetrachloro-6-(trichloromethyl)pyridine | 1.5 | | Zhang et al. (2010) | Q | 287, 288 |
| $C_6Cl_7N$ | $6.7\times10^{-2}$ | | Zhang et al. (2010) | Q | 287, 289 |
| [1134-04-9] | $7.0\times10^{-2}$ | | Zhang et al. (2010) | Q | 287, 290 |
| YMBFWRZKTZICHS-UHFFFAOYSA-N | $2.3\times10^{-1}$ | | Zhang et al. (2010) | Q | 287, 291 |
| 3,4,5,6-tetrachloropyridine-2-carbonitrile | 7.5 | | Zhang et al. (2010) | Q | 287, 288 |
| $C_6Cl_4N_2$ | 1.7 | | Zhang et al. (2010) | Q | 287, 289 |
| [17824-83-8] | $8.0\times10^{-1}$ | | Zhang et al. (2010) | Q | 287, 290 |
| KFPBGJYBKSQIAI-UHFFFAOYSA-N | $1.5\times10^{1}$ | | Zhang et al. (2010) | Q | 287, 291 |
| 4-amino-3,5,6-trichloropyridine-2-carbonitrile | $1.6\times10^{4}$ | | Zhang et al. (2010) | Q | 287, 288 |
| $C_6H_2Cl_3N_3$ | $1.1\times10^{3}$ | | Zhang et al. (2010) | Q | 287, 289 |
| [14143-60-3] | $9.0\times10^{3}$ | | Zhang et al. (2010) | Q | 287, 290 |
| AZYYQVGBVUCIEO-UHFFFAOYSA-N | $1.0\times10^{5}$ | | Zhang et al. (2010) | Q | 287, 291 |
| crimidine<br>$C_7H_{10}ClN_3$<br>[535-89-7]<br>HJIUPFPIEBPYIE-UHFFFAOYSA-N | $2.6\times10^{2}$ | | HSDB (2015) | Q | 99 |



Table A6.8: Chlorocarbons with nitrogen (C, H, O, N, Cl) (. . . continued)

| Substance Formula (Trivial Name) [CAS Registry Number] InChIKey | $H_s^{cp}$ (at $T^\ominus$) $\left[\dfrac{\mathrm{mol}}{\mathrm{m}^3\,\mathrm{Pa}}\right]$ | $\dfrac{\mathrm{d}\ln H_s^{cp}}{\mathrm{d}(1/T)}$ [K] | Reference | Type | Note |
|---|---|---|---|---|---|
| simazine $C_7H_{12}ClN_5$ [122-34-9] ODCWYMIRDDJXKW-UHFFFAOYSA-N | $1.0\times10^4$ | | HSDB (2015) | V | |
| | $2.9\times10^3$ | | Mackay et al. (2006d) | V | |
| | $2.9\times10^3$ | | Suntio et al. (1988) | V | 12 |
| | $1.6\times10^4$ | | Glotfelty et al. (1987) | V | |
| | $2.9\times10^1$ | | Barcelo and Hennion (1997) | X | 567 |
| | $6.2\times10^7$ | | Delgado and Alderete (2003) | C | |
| | $1.1\times10^4$ | | Delgado and Alderete (2003) | C | |
| | $9.1\times10^1$ | | Goodarzi et al. (2010) | Q | 568 |
| | $1.7\times10^3$ | | Hilal et al. (2008) | Q | |
| | $7.2\times10^3$ | | Abraham et al. (2007) | Q | |
| | $5.5\times10^5$ | | Delgado and Alderete (2003) | Q | |
| | $4.0\times10^6$ | | Delgado and Alderete (2003) | Q | |
| desethylterbuthylazine $C_7H_{12}ClN_5$ [30125-63-4] LMKQNTMFZLAJDV-UHFFFAOYSA-N | $2.2\times10^3$ | | Otto et al. (1997) | V | |
| mepiquat chloride $C_7H_{16}NCl$ [24307-26-4] VHOVSQVSAAQANU-UHFFFAOYSA-M | $3.3\times10^{11}$ | | Maniere et al. (2011) | ? | 241, 165 |
| atrazine $C_8H_{14}ClN_5$ [1912-24-9] MXWJVTOOROXGIU-UHFFFAOYSA-N | $1.9\times10^3$ | | Muir et al. (2004) | L | 367 |
| | $3.5\times10^3$ | | Mackay et al. (2006d) | V | |
| | $1.0\times10^3$ | | Siebers et al. (1994) | V | |
| | $3.3\times10^3$ | | Riederer (1990) | V | |
| | $3.4\times10^3$ | | Suntio et al. (1988) | V | 12 |
| | $2.0\times10^3$ | | Glotfelty et al. (1987) | V | |
| | $3.4\times10^1$ | | Barcelo and Hennion (1997) | X | 567 |
| | $8.3\times10^6$ | | Delgado and Alderete (2003) | C | |
| | $4.3\times10^3$ | | Delgado and Alderete (2003) | C | |
| | $7.2\times10^1$ | | Goodarzi et al. (2010) | Q | 568 |
| | $7.2\times10^2$ | | Hilal et al. (2008) | Q | |
| | $5.1\times10^3$ | | Abraham et al. (2007) | Q | |
| | $2.8\times10^4$ | | Delgado and Alderete (2003) | Q | |
| | $4.0\times10^5$ | | Delgado and Alderete (2003) | Q | |
| clonidine $C_9H_9Cl_2N_3$ [4205-90-7] GJSURZIOUXUGAL-UHFFFAOYSA-N | $6.6\times10^5$ | | HSDB (2015) | Q | 99 |
| cyprazine $C_9H_{12}ClN_5$ [22936-86-3] OOHIAOSLOGDBCE-UHFFFAOYSA-N | $7.6\times10^3$ | | Duchowicz et al. (2020) | V | 186 |
| | $7.7\times10^4$ | | Duchowicz et al. (2020) | Q | |
| | $3.8\times10^3$ | | HSDB (2015) | Q | 99 |



Table A6.8: Chlorocarbons with nitrogen (C, H, O, N, Cl) (. . . continued)

| Substance Formula (Trivial Name) [CAS Registry Number] InChIKey | $H_s^{cp}$ (at $T^{\ominus}$) $\left[\dfrac{\text{mol}}{\text{m}^3\,\text{Pa}}\right]$ | $\dfrac{\text{d}\ln H_s^{cp}}{\text{d}(1/T)}$ [K] | Reference | Type | Note |
|---|---|---|---|---|---|
| propazine C$_9$H$_{16}$ClN$_5$ [139-40-2] WJNRPILHGGKWCK-UHFFFAOYSA-N | $2.1\times10^3$ $1.0\times10^4$ $9.9\times10^1$ $1.7\times10^2$ $3.6\times10^2$ $4.0\times10^3$ | | HSDB (2015) Mackay et al. (2006d) Suntio et al. (1988) Barcelo and Hennion (1997) Goodarzi et al. (2010) Hilal et al. (2008) Abraham et al. (2007) | V V V X Q Q Q | 558 12 567 568, 571 |
| sebuthylazine C$_9$H$_{16}$ClN$_5$ [7286-69-3] BZRUVKZGXNSXMB-UHFFFAOYSA-N | $8.4\times10^2$ | | Ebert et al. (2023) | ? | 318 |
| terbuthylazine C$_9$H$_{16}$ClN$_5$ [5915-41-3] FZXISNSWEXTPMF-UHFFFAOYSA-N | $4.3\times10^2$ $2.5\times10^2$ $2.5\times10^2$ $2.4\times10^2$ $2.9\times10^2$ $9.0\times10^2$ $4.3\times10^2$ | | HSDB (2015) Mackay et al. (2006d) Otto et al. (1997) Siebers et al. (1994) Hilal et al. (2008) Abraham et al. (2007) Maniere et al. (2011) | V V V V Q Q ? | 241, 165 |
| cyanazine C$_9$H$_{13}$ClN$_6$ [21725-46-2] MZZBPDKVEFVLFF-UHFFFAOYSA-N | $3.3\times10^6$ $3.3\times10^4$ $8.3\times10^9$ $3.9\times10^6$ $9.6\times10^1$ $6.4\times10^5$ $2.0\times10^6$ $4.5\times10^6$ $1.0\times10^9$ | | Mackay et al. (2006d) Barcelo and Hennion (1997) Delgado and Alderete (2003) Delgado and Alderete (2003) Goodarzi et al. (2010) Hilal et al. (2008) Abraham et al. (2007) Delgado and Alderete (2003) Delgado and Alderete (2003) | V X C C Q Q Q Q Q | 567 568 |
| anilazine C$_9$H$_5$Cl$_3$N$_4$ [101-05-3] IMHBYKMAHXWHRP-UHFFFAOYSA-N | $3.5\times10^4$ $3.5\times10^4$ $2.9\times10^1$ $1.2\times10^3$ $9.5$ $5.4\times10^3$ $3.5\times10^4$ | | HSDB (2015) Mackay et al. (2006d) Zhang et al. (2010) Zhang et al. (2010) Zhang et al. (2010) Zhang et al. (2010) MacBean (2012a) | V V Q Q Q Q ? | 287, 288 287, 289 287, 290 287, 291 |
| 4,7-dichloroquinoline C$_9$H$_5$Cl$_2$N [86-98-6] HXEWMTXDBOQQKO-UHFFFAOYSA-N | $2.6\times10^1$ $6.9$ $1.5$ $7.0$ | | Zhang et al. (2010) Zhang et al. (2010) Zhang et al. (2010) Zhang et al. (2010) | Q Q Q Q | 287, 288 287, 289 287, 290 287, 291 |
| fenclorim C$_{10}$H$_6$Cl$_2$N$_2$ [3740-92-9] NRFQZTCQAYEXEE-UHFFFAOYSA-N | $5.6\times10^{-1}$ | | Ebert et al. (2023) | ? | 318 |



Table A6.8: Chlorocarbons with nitrogen (C, H, O, N, Cl) (... continued)

| Substance Formula (Trivial Name) [CAS Registry Number] InChIKey | $H_s^{cp}$ (at $T^\ominus$) $\left[\dfrac{\text{mol}}{\text{m}^3\,\text{Pa}}\right]$ | $\dfrac{\text{d}\ln H_s^{cp}}{\text{d}(1/T)}$ [K] | Reference | Type | Note |
|---|---|---|---|---|---|
| acetamiprid $C_{10}H_{11}ClN_4$ [135410-20-7] WCXDHFDTOYPNIE-UHFFFAOYSA-N | $1.4\times10^2$ $>1.9\times10^7$ | | HSDB (2015) Maniere et al. (2011) | Q ? | 99 165 |
| pyrimethamine $C_{12}H_{13}ClN_4$ [58-14-0] WKSAUQYGYAYLPV-UHFFFAOYSA-N | $9.1\times10^4$ | | HSDB (2015) | Q | 447 |
| penconazole $C_{13}H_{15}Cl_2N_3$ [66246-88-6] WKBPZYKAUNRMKP-UHFFFAOYSA-N | $6.9\times10^2$ $1.2\times10^3$ 2.1 $1.5\times10^3$ | | Duchowicz et al. (2020) Mackay et al. (2006d) Duchowicz et al. (2020) Maniere et al. (2011) | V V Q ? | 186 12, 165 |
| myclobutanil $C_{15}H_{17}ClN_4$ [88671-89-0] HZJKXKUJVSEEFU-UHFFFAOYSA-N | $2.3\times10^3$ $2.3\times10^3$ $2.2\times10^2$ $2.3\times10^3$ | | Duchowicz et al. (2020) HSDB (2015) Duchowicz et al. (2020) Maniere et al. (2011) | V V Q ? | 186 12, 165 |
| 2-chloro-4,6-bis(2,4-dimethylphenyl)-1,3,5-triazine $C_{19}H_{18}N_3Cl$ [1237-53-2] LVWOBZPDFCTAOU-UHFFFAOYSA-N | $1.2\times10^3$ $1.3\times10^3$ 7.0 $1.2\times10^3$ | | Zhang et al. (2010) Zhang et al. (2010) Zhang et al. (2010) Zhang et al. (2010) | Q Q Q Q | 287, 288 287, 289 287, 290 287, 291 |
| fenbuconazole $C_{19}H_{17}ClN_4$ [114369-43-6] RQDJADAKIFFEKQ-UHFFFAOYSA-N | $3.3\times10^4$ | | Maniere et al. (2011) | ? | 241, 165 |
| trichloronitromethane $CCl_3NO_2$ (chloropicrin) [76-06-2] LFHISGNCFUNFFM-UHFFFAOYSA-N | $4.7\times10^{-3}$ $4.7\times10^{-3}$ $4.7\times10^{-3}$ $4.7\times10^{-3}$ $4.0\times10^{-3}$ $4.8\times10^{-3}$ $5.1\times10^{-3}$ $5.0\times10^{-2}$ $2.6\times10^{-2}$ $2.5\times10^{-3}$ $3.0\times10^{-2}$ $4.8\times10^{-3}$ $1.3\times10^{-2}$ $4.8\times10^{-3}$ | | Burkholder et al. (2019) Burkholder et al. (2015) Sander et al. (2011) Worthington and Wade (2007) Welke et al. (1998) Kawamoto and Urano (1989) Mackay et al. (2006d) Suntio et al. (1988) Keshavarz et al. (2022) Duchowicz et al. (2020) Hilal et al. (2008) Modarresi et al. (2007) Yaffe et al. (2003) Katritzky et al. (1998) Duchowicz et al. (2020) | L L L M M M V V Q Q Q Q Q Q ? | 558 12 184 67 248, 249 185, 21 |



Table A6.8: Chlorocarbons with nitrogen (C, H, O, N, Cl) (...continued)

| Substance<br>Formula<br>(Trivial Name)<br>[CAS Registry Number]<br>InChIKey | $H_s^{cp}$<br>(at $T^{\ominus}$)<br>$\left[\dfrac{\mathrm{mol}}{\mathrm{m^3\,Pa}}\right]$ | $\dfrac{\mathrm{d}\ln H_s^{cp}}{\mathrm{d}(1/T)}$<br><br>[K] | Reference | Type | Note |
|---|---|---|---|---|---|
| phosgene oxime<br>CHCl$_2$NO<br>[1794-86-1]<br>JIRJHEXNDQBKRZ-UHFFFAOYSA-N | $1.8\times10^1$ | | HSDB (2015) | Q | 99 |
| 1,1-dichloro-1-nitroethane<br>C$_2$H$_3$Cl$_2$NO$_2$<br>[594-72-9]<br>OQOGEOLRYAOSKO-UHFFFAOYSA-N | $7.7\times10^{-3}$<br>$1.3\times10^{-1}$<br>$7.7\times10^{-3}$<br>$2.0\times10^{-2}$<br>$1.1\times10^{-1}$ | | Duchowicz et al. (2020)<br>Duchowicz et al. (2020)<br>HSDB (2015)<br>Hilal et al. (2008)<br>Modarresi et al. (2007) | V<br>Q<br>Q<br>Q<br>Q | 186<br><br>99<br><br>67 |
| 2-chloroacetamide<br>C$_2$H$_4$ClNO<br>[79-07-2]<br>VXIVSQZSERGHQP-UHFFFAOYSA-N | $2.5\times10^3$ | | HSDB (2015) | Q | 99 |
| MCM:CCL3PAN<br>C$_2$NO$_5$Cl$_3$<br>LLOMRHYBAIQQGH-UHFFFAOYSA-N | $1.0\times10^1$<br>$7.3\times10^{-1}$<br>$3.6\times10^{-5}$ | | Wang et al. (2017)<br>Wang et al. (2017)<br>Wang et al. (2017) | Q<br>Q<br>Q | 80, 238<br>80, 239<br>80, 240 |
| MCM:CCLNO3COOH<br>C$_2$H$_4$NO$_5$Cl<br>GVGGZHVOOFUNFY-UHFFFAOYSA-N | $2.1\times10^3$<br>$2.6\times10^3$<br>4.2 | | Wang et al. (2017)<br>Wang et al. (2017)<br>Wang et al. (2017) | Q<br>Q<br>Q | 80, 238<br>80, 239<br>80, 240 |
| MCM:CHCL2PAN<br>C$_2$HNO$_5$Cl$_2$<br>BORKLXDOXKHVRO-UHFFFAOYSA-N | $2.2\times10^1$<br>$1.6\times10^1$<br>$4.9\times10^{-4}$ | | Wang et al. (2017)<br>Wang et al. (2017)<br>Wang et al. (2017) | Q<br>Q<br>Q | 80, 238<br>80, 239<br>80, 240 |
| MCM:CLETPAN<br>C$_2$H$_2$NO$_5$Cl<br>IEQWXJNETVFAEB-UHFFFAOYSA-N | 9.8<br>$3.3\times10^1$<br>$4.2\times10^{-3}$ | | Wang et al. (2017)<br>Wang et al. (2017)<br>Wang et al. (2017) | Q<br>Q<br>Q | 80, 238<br>80, 239<br>80, 240 |
| MCM:CNO3CLOOH<br>C$_2$H$_4$NO$_5$Cl<br>FBIXZLOOAAICML-UHFFFAOYSA-N | $2.1\times10^3$<br>$3.0\times10^3$<br>8.3 | | Wang et al. (2017)<br>Wang et al. (2017)<br>Wang et al. (2017) | Q<br>Q<br>Q | 80, 238<br>80, 239<br>80, 240 |
| MCM:CCLNO3COH<br>C$_2$H$_4$NO$_4$Cl<br>PMLDKFZZOPGPNK-UHFFFAOYSA-N | $1.8\times10^2$<br>$7.1\times10^2$<br>$7.8\times10^{-1}$ | | Wang et al. (2017)<br>Wang et al. (2017)<br>Wang et al. (2017) | Q<br>Q<br>Q | 80, 238<br>80, 239<br>80, 240 |
| MCM:CL2OHPAN<br>C$_2$HNO$_6$Cl$_2$<br>YJZUCQDIDXVPNP-UHFFFAOYSA-N | $3.8\times10^3$<br>$1.5\times10^2$<br>$1.4\times10^{-2}$ | | Wang et al. (2017)<br>Wang et al. (2017)<br>Wang et al. (2017) | Q<br>Q<br>Q | 80, 238<br>80, 239<br>80, 240 |
| MCM:CLOHPAN<br>C$_2$H$_2$NO$_6$Cl<br>ZNYKQWDXHYNGTP-UHFFFAOYSA-N | $4.2\times10^3$<br>$1.0\times10^4$<br>$8.9\times10^{-2}$ | | Wang et al. (2017)<br>Wang et al. (2017)<br>Wang et al. (2017) | Q<br>Q<br>Q | 80, 238<br>80, 239<br>80, 240 |
| 1-chloro-1-nitroethane<br>C$_2$H$_4$ClNO$_2$<br>[598-92-5]<br>LPIWIOBGUAPNQW-UHFFFAOYSA-N | $2.0\times10^{-2}$ | | Ebert et al. (2023) | ? | 316 |



Table A6.8: Chlorocarbons with nitrogen (C, H, O, N, Cl) (... continued)

| Substance Formula (Trivial Name) [CAS Registry Number] InChIKey | $H_s^{cp}$ (at $T^{\ominus}$) $\left[\dfrac{\mathrm{mol}}{\mathrm{m}^3\,\mathrm{Pa}}\right]$ | $\dfrac{\mathrm{d}\ln H_s^{cp}}{\mathrm{d}(1/T)}$ [K] | Reference | Type | Note |
|---|---|---|---|---|---|
| MCM:CNO3CLOH $C_2H_4NO_4Cl$ OUFVLIWCNZOATK-UHFFFAOYSA-N | $2.8\times10^2$ $1.4\times10^3$ $1.4\times10^2$ | | Wang et al. (2017) Wang et al. (2017) Wang et al. (2017) | Q Q Q | 80, 238 80, 239 80, 240 |
| MCM:CCLCONO3 $C_2H_2NO_4Cl$ XDEFPIDZLAIVCM-UHFFFAOYSA-N | $2.6\times10^1$ $1.6\times10^2$ $2.5\times10^{-3}$ | | Wang et al. (2017) Wang et al. (2017) Wang et al. (2017) | Q Q Q | 80, 238 80, 239 80, 240 |
| MCM:CNO3OCL $C_2H_2NO_4Cl$ ODQFUUDIEKKDDJ-UHFFFAOYSA-N | $1.7\times10^1$ $1.1\times10^1$ $4.4\times10^{-3}$ | | Wang et al. (2017) Wang et al. (2017) Wang et al. (2017) | Q Q Q | 80, 238 80, 239 80, 240 |
| symclosene $C_3Cl_3N_3O_3$ (trichloroisocyanuric acid) [87-90-1] YRIZYWQGELRKNT-UHFFFAOYSA-N | $1.6\times10^5$ | | HSDB (2015) | Q | 99 |
| MCM:CL12PAN $C_3H_3NO_5Cl_2$ MSMFQRZPTBDNBF-UHFFFAOYSA-N | $6.2\times10^1$ $4.9\times10^1$ $2.5\times10^{-3}$ | | Wang et al. (2017) Wang et al. (2017) Wang et al. (2017) | Q Q Q | 80, 238 80, 239 80, 240 |
| 1-chloro-1-nitropropane $C_3H_6ClNO_2$ [600-25-9] XEKUXTMJEFPWCG-UHFFFAOYSA-N | $4.6\times10^{-2}$ | | Ebert et al. (2023) | ? | 316 |
| 1,3-dichloro-5,5-dimethylhydantoin $C_5H_6Cl_2N_2O_2$ [118-52-5] KEQGZUUPPQEDPF-UHFFFAOYSA-N | 9.9 | | HSDB (2015) | Q | 99 |
| carmustine $C_5H_9Cl_2N_3O_2$ [154-93-8] DLGOEMSEDOSKAD-UHFFFAOYSA-N | $2.1\times10^5$ | | HSDB (2015) | Q | 99 |
| 2-chloro-N,N-di-2-propenylacetamide $C_8H_{12}ClNO$ [93-71-0] MDBGGTQNNUOQRC-UHFFFAOYSA-N | $9.2\times10^1$ $9.2\times10^1$ 9.5 $9.7\times10^1$ $1.9\times10^1$ | | Duchowicz et al. (2020) HSDB (2015) Duchowicz et al. (2020) Hilal et al. (2008) Modarresi et al. (2007) | V V Q Q Q | 186    67 |
| 2,2-dichloro-N,N-di-2-propenylacetamide $C_8H_{11}Cl_2NO$ (dichlormid) [37764-25-3] YRMLFORXOOIJDR-UHFFFAOYSA-N | $3.1\times10^1$ $2.7\times10^1$ | | Hilal et al. (2008) Modarresi et al. (2007) | Q Q | 67 |



Table A6.8: Chlorocarbons with nitrogen (C, H, O, N, Cl) (...continued)

| Substance<br>Formula<br>(Trivial Name)<br>[CAS Registry Number]<br>InChIKey | $H_s^{cp}$<br>(at $T^\ominus$)<br>$\left[\dfrac{\mathrm{mol}}{\mathrm{m^3\,Pa}}\right]$ | $\dfrac{\mathrm{d}\ln H_s^{cp}}{\mathrm{d}(1/T)}$<br><br>[K] | Reference | Type | Note |
|---|---|---|---|---|---|
| lomustine<br>$C_9H_{16}ClN_3O_2$<br>[13010-47-4]<br>GQYIWUVLTXOXAJ-UHFFFAOYSA-N | $5.5\times10^4$ | | HSDB (2015) | Q | 99 |
| semustine<br>$C_{10}H_{18}ClN_3O_2$<br>[13909-09-6]<br>FVLVBPDQNARYJU-UHFFFAOYSA-N | $3.9\times10^4$ | | HSDB (2015) | Q | 99 |
| furilazole<br>$C_{11}H_{13}Cl_2NO_3$<br>[121776-33-8]<br>MCNOFYBITGAAGM-UHFFFAOYSA-N | $1.1\times10^5$<br>$1.3\times10^3$ | | Duchowicz et al. (2020)<br>Duchowicz et al. (2020) | V<br>Q | 186 |
| dimethazone<br>$C_{12}H_{14}ClNO_2$<br>(clomazone)<br>[81777-89-1]<br>KIEDNEWSYUYDSN-UHFFFAOYSA-N | $2.4\times10^2$<br>$2.4\times10^2$<br>4.5<br>$2.4\times10^2$ | | Duchowicz et al. (2020)<br>MacBean (2012b)<br>Duchowicz et al. (2020)<br>Maniere et al. (2011) | V<br>X<br>Q<br>? | 186<br>350<br><br>241, 165 |
| 2-chloronitrobenzene<br>$C_6H_4ClNO_2$<br>(o-chloronitrobenzene)<br>[88-73-3]<br>BFCFYVKQTRLZHA-UHFFFAOYSA-N | 1.1<br>$2.2\times10^{-1}$<br>$2.8\times10^{-1}$<br>1.1<br>3.1<br>$6.2\times10^{-1}$<br>$1.5\times10^{-1}$<br>1.2<br>$4.6\times10^{-1}$<br>$3.1\times10^{-1}$<br>$1.8\times10^{-1}$<br><br>$2.3\times10^{-1}$<br>$8.1\times10^{-1}$<br>$7.5\times10^{-1}$<br>1.1<br><br>$9.7\times10^{-1}$ | <br><br><br><br><br><br><br><br><br><br><br>4700<br><br><br><br><br>6000 | Altschuh et al. (1999)<br>Hellmann (1987)<br>Lide and Frederikse (1995)<br>Keshavarz et al. (2022)<br>Duchowicz et al. (2020)<br>Zhang et al. (2010)<br>Zhang et al. (2010)<br>Zhang et al. (2010)<br>Zhang et al. (2010)<br>Hilal et al. (2008)<br>Modarresi et al. (2007)<br>Kühne et al. (2005)<br>Yaffe et al. (2003)<br>Yao et al. (2002)<br>Katritzky et al. (1998)<br>Duchowicz et al. (2020)<br>Kühne et al. (2005)<br>Yaws (1999) | M<br>M<br>V<br>Q<br>Q<br>Q<br>Q<br>Q<br>Q<br>Q<br>Q<br>Q<br>Q<br>Q<br>Q<br>?<br>?<br>? | <br>87<br><br><br>184<br>287, 288<br>287, 289<br>287, 290<br>287, 291<br><br>67<br><br>248, 249<br>229<br><br>185, 21<br><br>21, 12 |
| 3-chloronitrobenzene<br>$C_6H_4ClNO_2$<br>(m-chloronitrobenzene)<br>[121-73-3]<br>KMAQZIILEGKYQZ-UHFFFAOYSA-N | 1.4<br>$7.3\times10^{-1}$<br>$1.1\times10^{-1}$<br>1.1<br>1.8<br>$6.2\times10^{-1}$<br>$2.8\times10^{-1}$<br>$2.8\times10^{-1}$<br>$4.6\times10^{-1}$<br>$2.1\times10^{-1}$ | | Chao et al. (2017)<br>Altschuh et al. (1999)<br>Schüürmann (2000)<br>Keshavarz et al. (2022)<br>Duchowicz et al. (2020)<br>Zhang et al. (2010)<br>Zhang et al. (2010)<br>Zhang et al. (2010)<br>Zhang et al. (2010)<br>Hilal et al. (2008) | M<br>M<br>V<br>Q<br>Q<br>Q<br>Q<br>Q<br>Q<br>Q | <br><br><br><br><br>287, 288<br>287, 289<br>287, 290<br>287, 291<br> |



Table A6.8: Chlorocarbons with nitrogen (C, H, O, N, Cl) (...continued)

| Substance<br>Formula<br>(Trivial Name)<br>[CAS Registry Number]<br>InChIKey | $H_s^{cp}$<br>(at $T^\ominus$)<br>$\left[\dfrac{\mathrm{mol}}{\mathrm{m^3\,Pa}}\right]$ | $\dfrac{\mathrm{d}\ln H_s^{cp}}{\mathrm{d}(1/T)}$<br><br>[K] | Reference | Type | Note |
|---|---|---|---|---|---|
| | $1.7\times10^{-1}$ | | Modarresi et al. (2007) | Q | 67 |
| | $7.3\times10^{-1}$ | | Duchowicz et al. (2020) | ? | 185, 21 |
| | $3.4\times10^{-1}$ | | Yaws (1999) | ? | 21, 12 |
| 4-chloronitrobenzene | $6.4\times10^{-1}$ | 5900 | Brockbank (2013) | L | 1 |
| $C_6H_4ClNO_2$ | 1.4 | | Chao et al. (2017) | M | |
| (*p*-chloronitrobenzene) | 2.0 | | Altschuh et al. (1999) | M | |
| [100-00-5] | $1.8\times10^{-1}$ | | Hellmann (1987) | M | 87 |
| CZGCEKJOLUNIFY-UHFFFAOYSA-N | $2.8\times10^{-1}$ | | Lide and Frederikse (1995) | V | |
| | 1.1 | | Keshavarz et al. (2022) | Q | |
| | 1.7 | | Duchowicz et al. (2020) | Q | 184 |
| | $6.2\times10^{-1}$ | | Zhang et al. (2010) | Q | 287, 288 |
| | $3.0\times10^{-1}$ | | Zhang et al. (2010) | Q | 287, 289 |
| | $6.1\times10^{-1}$ | | Zhang et al. (2010) | Q | 287, 290 |
| | $4.6\times10^{-1}$ | | Zhang et al. (2010) | Q | 287, 291 |
| | $2.3\times10^{-1}$ | | Hilal et al. (2008) | Q | |
| | $2.3\times10^{-1}$ | | Hilal et al. (2008) | Q | |
| | $1.4\times10^{-1}$ | | Modarresi et al. (2007) | Q | 67 |
| | | 4700 | Kühne et al. (2005) | Q | |
| | $1.9\times10^{-1}$ | | Yaffe et al. (2003) | Q | 248, 249 |
| | 1.6 | | Katritzky et al. (1998) | Q | |
| | 2.0 | | Duchowicz et al. (2020) | ? | 185, 21 |
| | | 4000 | Kühne et al. (2005) | ? | |
| 1,2-dichloro-4-nitrobenzene | 1.2 | | Altschuh et al. (1999) | M | |
| $C_6H_3Cl_2NO_2$ | 1.1 | | Keshavarz et al. (2022) | Q | |
| [99-54-7] | 1.8 | | Duchowicz et al. (2020) | Q | |
| NTBYINQTYWZXLH-UHFFFAOYSA-N | $2.1\times10^{-1}$ | | Abraham et al. (2019) | Q | |
| | $8.4\times10^{-1}$ | | Zhang et al. (2010) | Q | 287, 288 |
| | $3.1\times10^{-1}$ | | Zhang et al. (2010) | Q | 287, 289 |
| | $4.6\times10^{-1}$ | | Zhang et al. (2010) | Q | 287, 290 |
| | $6.7\times10^{-1}$ | | Zhang et al. (2010) | Q | 287, 291 |
| | $2.7\times10^{-1}$ | | Hilal et al. (2008) | Q | |
| | $2.3\times10^{-1}$ | | Modarresi et al. (2007) | Q | 67 |
| | 1.2 | | Duchowicz et al. (2020) | ? | 185, 21 |
| 1,4-dichloro-2-nitrobenzene | $8.2\times10^{-1}$ | | Altschuh et al. (1999) | M | |
| $C_6H_3Cl_2NO_2$ | 1.1 | | Keshavarz et al. (2022) | Q | |
| [89-61-2] | 1.8 | | Duchowicz et al. (2020) | Q | 184 |
| RZKKOBGFCAHLCZ-UHFFFAOYSA-N | $3.2\times10^{-1}$ | | Abraham et al. (2019) | Q | |
| | $8.4\times10^{-1}$ | | Zhang et al. (2010) | Q | 287, 288 |
| | $1.5\times10^{-1}$ | | Zhang et al. (2010) | Q | 287, 289 |
| | $7.3\times10^{-1}$ | | Zhang et al. (2010) | Q | 287, 290 |
| | $6.1\times10^{-1}$ | | Zhang et al. (2010) | Q | 287, 291 |
| | $3.1\times10^{-1}$ | | Hilal et al. (2008) | Q | |
| | $2.2\times10^{-1}$ | | Modarresi et al. (2007) | Q | 67 |
| | $8.2\times10^{-1}$ | | Duchowicz et al. (2020) | ? | 185, 21 |





Table A6.8: Chlorocarbons with nitrogen (C, H, O, N, Cl) (. . . continued)

| Substance Formula (Trivial Name) [CAS Registry Number] InChIKey | $H_s^{cp}$ (at $T^\ominus$) $\left[\dfrac{\text{mol}}{\text{m}^3\,\text{Pa}}\right]$ | $\dfrac{\text{d}\ln H_s^{cp}}{\text{d}(1/T)}$ [K] | Reference | Type | Note |
|---|---|---|---|---|---|
| 2,3-dichloronitrobenzene | $5.0\times10^{-1}$ | | Abraham et al. (2019) | Q | |
| $C_6H_3Cl_2NO_2$ | $8.2\times10^{-1}$ | | HSDB (2015) | Q | 99 |
| [3209-22-1] | $8.4\times10^{-1}$ | | Zhang et al. (2010) | Q | 287, 288 |
| CMVQZRLQEOAYSW-UHFFFAOYSA-N | $1.4\times10^{-1}$ | | Zhang et al. (2010) | Q | 287, 289 |
| | $9.7\times10^{-1}$ | | Zhang et al. (2010) | Q | 287, 290 |
| | $6.7\times10^{-1}$ | | Zhang et al. (2010) | Q | 287, 291 |
| 2,4-dichloronitrobenzene | $3.1\times10^{-1}$ | | HSDB (2015) | Q | 99 |
| $C_6H_3Cl_2NO_2$ | $8.4\times10^{-1}$ | | Zhang et al. (2010) | Q | 287, 288 |
| [611-06-3] | $1.6\times10^{-1}$ | | Zhang et al. (2010) | Q | 287, 289 |
| QUIMTLZDMCNYGY-UHFFFAOYSA-N | $8.4\times10^{-1}$ | | Zhang et al. (2010) | Q | 287, 290 |
| | $2.9\times10^{-1}$ | | Zhang et al. (2010) | Q | 287, 291 |
| 3,5-dichloronitrobenzene | $8.4\times10^{-1}$ | | Zhang et al. (2010) | Q | 287, 288 |
| $C_6H_3Cl_2NO_2$ | $2.0\times10^{-1}$ | | Zhang et al. (2010) | Q | 287, 289 |
| [618-62-2] | $1.1\times10^{-1}$ | | Zhang et al. (2010) | Q | 287, 290 |
| RNABGKOKSBUFHW-UHFFFAOYSA-N | $2.9\times10^{-1}$ | | Zhang et al. (2010) | Q | 287, 291 |
| pentachloronitrobenzene | $2.7$ | | Kawamoto and Urano (1989) | M | |
| $C_6Cl_5NO_2$ | $2.2\times10^{-1}$ | | Duchowicz et al. (2020) | V | 186 |
| (quintozene) | $2.2\times10^{-1}$ | | HSDB (2015) | V | |
| [82-68-8] | $2.3\times10^{-1}$ | | Mackay et al. (2006d) | V | |
| LKPLKUMXSAEKID-UHFFFAOYSA-N | $2.1\times10^{-1}$ | | Howard and Meylan (1997) | X | 446 |
| | $1.1$ | | Duchowicz et al. (2020) | Q | |
| | $2.1$ | | Zhang et al. (2010) | Q | 287, 288 |
| | $2.3\times10^{-2}$ | | Zhang et al. (2010) | Q | 287, 289 |
| | $2.2\times10^{-2}$ | | Zhang et al. (2010) | Q | 287, 290 |
| | $2.2\times10^{-1}$ | | Zhang et al. (2010) | Q | 287, 291 |
| | $6.9\times10^{-2}$ | | Hilal et al. (2008) | Q | |
| | $2.1$ | | Meylan and Howard (1991) | Q | |
| 4-chloro-2-nitrophenol | $7.8\times10^{-1}$ | | Duchowicz et al. (2020) | V | 186 |
| $C_6H_4ClNO_3$ | $7.8\times10^{-1}$ | | Schwarzenbach et al. (1988) | V | 12 |
| [89-64-5] | $1.6\times10^{1}$ | | Duchowicz et al. (2020) | Q | |
| NWSIFTLPLKCTSX-UHFFFAOYSA-N | $7.9\times10^{-1}$ | | Yaffe et al. (2003) | Q | 248, 249 |
| | $3.6\times10^{1}$ | | Katritzky et al. (1998) | Q | |
| 2-chloro-4-nitrobenzenamine | $1.0\times10^{3}$ | | Altschuh et al. (1999) | M | |
| $C_6H_5ClN_2O_2$ | $4.2\times10^{3}$ | | Keshavarz et al. (2022) | Q | |
| [121-87-9] | $1.3\times10^{3}$ | | Duchowicz et al. (2020) | Q | 184 |
| LOCWBQIWHWIRGN-UHFFFAOYSA-N | $1.8\times10^{3}$ | | Zhang et al. (2010) | Q | 287, 288 |
| | $1.3\times10^{3}$ | | Zhang et al. (2010) | Q | 287, 289 |
| | $6.7\times10^{4}$ | | Zhang et al. (2010) | Q | 287, 290 |
| | $2.1\times10^{3}$ | | Zhang et al. (2010) | Q | 287, 291 |
| | $4.6\times10^{2}$ | | Hilal et al. (2008) | Q | |
| | $6.4\times10^{2}$ | | Modarresi et al. (2007) | Q | 67 |
| | $1.0\times10^{3}$ | | Duchowicz et al. (2020) | ? | 185, 21 |



Table A6.8: Chlorocarbons with nitrogen (C, H, O, N, Cl) (. . . continued)

| Substance Formula (Trivial Name) [CAS Registry Number] InChIKey | $H_s^{cp}$ (at $T^{\ominus}$) $\left[\dfrac{\text{mol}}{\text{m}^3\,\text{Pa}}\right]$ | $\dfrac{\text{d}\ln H_s^{cp}}{\text{d}(1/T)}$ [K] | Reference | Type | Note |
|---|---|---|---|---|---|
| 2-chloro-5-nitrobenzenamine $C_6H_5ClN_2O_2$ [6283-25-6] KWIXNFOTNVKIGM-UHFFFAOYSA-N | $1.8\times10^3$ | | HSDB (2015) | Q | 99 |
| 4-chloro-2,6-dinitrobenzenamine $C_6H_4ClN_3O_4$ [5388-62-5] CLMQUEQFVUMDPC-UHFFFAOYSA-N | $7.6\times10^1$ | | HSDB (2015) | Q | 99 |
| 1-chloro-2,4-dinitrobenzene $C_6H_3ClN_2O_4$ [97-00-7] VYZAHLCBVHPDDF-UHFFFAOYSA-N | 3.5 4.0 $1.1\times10^1$ $1.8\times10^2$ $1.6\times10^2$ 6.0 5.3 $3.9\times10^1$ $1.0\times10^1$ | | Duchowicz et al. (2020) HSDB (2015) Yaws (2003) Duchowicz et al. (2020) Zhang et al. (2010) Zhang et al. (2010) Zhang et al. (2010) Zhang et al. (2010) Gharagheizi et al. (2010) | V V X Q Q Q Q Q Q | 186 <br> 237, 80 <br> 287, 288 287, 289 287, 290 287, 291 246 |
| 1-chloro-2,6-dinitrobenzene $C_6H_3ClN_2O_4$ [606-21-3] BPPMIQPXQVIZNJ-UHFFFAOYSA-N | $1.6\times10^2$ 4.3 7.2 $3.9\times10^1$ | | Zhang et al. (2010) Zhang et al. (2010) Zhang et al. (2010) Zhang et al. (2010) | Q Q Q Q | 287, 288 287, 289 287, 290 287, 291 |
| 2-chloro-1,3,5-trinitrobenzene $C_6H_2ClN_3O_6$ [88-88-0] HJRJRUMKQCMYDL-UHFFFAOYSA-N | $3.9\times10^4$ | | HSDB (2015) | Q | 99 |
| 2,3,4-trichloronitrobenzene $C_6H_2Cl_3NO_2$ [17700-09-3] BGKIECJVXXHLDP-UHFFFAOYSA-N | 1.1 $1.3\times10^{-1}$ $2.0\times10^{-1}$ $4.0\times10^{-1}$ | | Zhang et al. (2010) Zhang et al. (2010) Zhang et al. (2010) Zhang et al. (2010) | Q Q Q Q | 287, 288 287, 289 287, 290 287, 291 |
| 2,3,4,5-tetrachloronitrobenzene $C_6HCl_4NO_2$ [879-39-0] MTBYTWZDRVOMBR-UHFFFAOYSA-N | $4.3\times10^{-1}$ | | HSDB (2015) | Q | 99 |
| 1,2,4,5-tetrachloronitrobenzene $C_6HCl_4NO_2$ (tecnazene) [117-18-0] XQTLDIFVVHJORV-UHFFFAOYSA-N | $4.3\times10^{-1}$ | | HSDB (2015) | Q | 99 |
| 4-chloro-2-nitrobenzenamine $C_6H_5ClN_2O_2$ [89-63-4] PBGKNXWGYQPUJK-UHFFFAOYSA-N | $8.2\times10^1$ $8.0\times10^1$ $1.7\times10^2$ $2.2\times10^3$ $2.9\times10^2$ | | HSDB (2015) Zhang et al. (2010) Zhang et al. (2010) Zhang et al. (2010) Zhang et al. (2010) | Q Q Q Q Q | 99 287, 288 287, 289 287, 290 287, 291 |



Table A6.8: Chlorocarbons with nitrogen (C, H, O, N, Cl) (...continued)

| Substance<br>Formula<br>(Trivial Name)<br>[CAS Registry Number]<br>InChIKey | $H_s^{cp}$<br>(at $T^\ominus$)<br>$\left[\dfrac{\text{mol}}{\text{m}^3\,\text{Pa}}\right]$ | $\dfrac{\mathrm{d}\ln H_s^{cp}}{\mathrm{d}(1/T)}$<br><br>[K] | Reference | Type | Note |
|---|---|---|---|---|---|
| 4-chloro-3-nitrobenzenamine<br>$C_6H_5ClN_2O_2$<br>[635-22-3]<br>FOHHWGVAOVDVLP-UHFFFAOYSA-N | $1.8\times10^3$<br>$1.8\times10^3$<br>$1.1\times10^3$<br>$2.4\times10^3$<br>$3.5\times10^3$ | | HSDB (2015)<br>Zhang et al. (2010)<br>Zhang et al. (2010)<br>Zhang et al. (2010)<br>Zhang et al. (2010) | Q<br>Q<br>Q<br>Q<br>Q | 99<br>287, 288<br>287, 289<br>287, 290<br>287, 291 |
| botran<br>$C_6H_4Cl_2N_2O_2$<br>[99-30-9]<br>BIXZHMJUSMUDOQ-UHFFFAOYSA-N | $1.2\times10^2$<br>$2.4\times10^3$<br>$6.9\times10^1$<br>$1.7\times10^3$<br>$1.4\times10^3$ | | HSDB (2015)<br>Zhang et al. (2010)<br>Zhang et al. (2010)<br>Zhang et al. (2010)<br>Zhang et al. (2010) | V<br>Q<br>Q<br>Q<br>Q | <br>287, 288<br>287, 289<br>287, 290<br>287, 291 |
| 3,5-dichlorophenyl isocyanate<br>$C_7H_3Cl_2NO$<br>[34893-92-0]<br>XEFUJGURFLOFAN-UHFFFAOYSA-N | $7.7\times10^{-2}$<br>$2.8$<br>$8.2\times10^{-3}$<br>$7.2\times10^{-1}$ | | Zhang et al. (2010)<br>Zhang et al. (2010)<br>Zhang et al. (2010)<br>Zhang et al. (2010) | Q<br>Q<br>Q<br>Q | 287, 288<br>287, 289<br>287, 290<br>287, 291 |
| 2-chloro-5-nitrobenzoic acid<br>$C_7H_4ClNO_4$<br>[2516-96-3]<br>QUEKGYQTRJVEQC-UHFFFAOYSA-N | $6.5\times10^3$ | | Abraham et al. (2019) | Q | |
| 4-chloro-3-nitrobenzoic acid<br>$C_7H_4ClNO_4$<br>[96-99-1]<br>DFXQXFGFOLXAPO-UHFFFAOYSA-N | $3.6\times10^3$<br>$3.1\times10^4$<br>$2.1\times10^3$<br>$1.6\times10^3$<br>$9.2\times10^4$ | | Abraham et al. (2019)<br>Zhang et al. (2010)<br>Zhang et al. (2010)<br>Zhang et al. (2010)<br>Zhang et al. (2010) | Q<br>Q<br>Q<br>Q<br>Q | <br>287, 288<br>287, 289<br>287, 290<br>287, 291 |
| 1-chloro-4-isocyanatobenzene<br>$C_7H_4ClNO$<br>[104-12-1]<br>ADAKRBAJFHTIEW-UHFFFAOYSA-N | $5.7\times10^{-2}$<br>$4.1$<br>$9.7\times10^{-3}$<br>$3.4$ | | Zhang et al. (2010)<br>Zhang et al. (2010)<br>Zhang et al. (2010)<br>Zhang et al. (2010) | Q<br>Q<br>Q<br>Q | 287, 288<br>287, 289<br>287, 290<br>287, 291 |
| 1,2-dichloro-4-isocyanatobenzene<br>$C_7H_3Cl_2NO$<br>[102-36-3]<br>MFUVCHZWGSJKEQ-UHFFFAOYSA-N | $7.7\times10^{-2}$<br>$4.5$<br>$1.6\times10^{-2}$<br>$2.2$ | | Zhang et al. (2010)<br>Zhang et al. (2010)<br>Zhang et al. (2010)<br>Zhang et al. (2010) | Q<br>Q<br>Q<br>Q | 287, 288<br>287, 289<br>287, 290<br>287, 291 |
| 2-chloro-1-methyl-4-nitrobenzene<br>$C_7H_6ClNO_2$<br>[121-86-8]<br>LLYXJBROWQDVMI-UHFFFAOYSA-N | $2.4\times10^{-1}$<br>$5.7\times10^{-1}$<br>$3.4\times10^{-1}$<br>$3.7\times10^{-1}$<br>$2.5\times10^{-1}$ | | HSDB (2015)<br>Zhang et al. (2010)<br>Zhang et al. (2010)<br>Zhang et al. (2010)<br>Zhang et al. (2010) | Q<br>Q<br>Q<br>Q<br>Q | 545<br>287, 288<br>287, 289<br>287, 290<br>287, 291 |
| 2,4-dichloro-3-methyl-6-nitrophenol<br>$C_7H_5Cl_2NO_3$<br>[39549-27-4]<br>VMBRJHMTAZXHES-UHFFFAOYSA-N | $2.3$<br>$2.9$<br>$9.5\times10^1$<br>$3.3\times10^{-1}$ | | Zhang et al. (2010)<br>Zhang et al. (2010)<br>Zhang et al. (2010)<br>Zhang et al. (2010) | Q<br>Q<br>Q<br>Q | 287, 288<br>287, 289<br>287, 290<br>287, 291 |



Table A6.8: Chlorocarbons with nitrogen (C, H, O, N, Cl) (...continued)

| Substance Formula (Trivial Name) [CAS Registry Number] InChIKey | $H_s^{cp}$ (at $T^\ominus$) $\left[ \dfrac{\mathrm{mol}}{\mathrm{m}^3\,\mathrm{Pa}} \right]$ | $\dfrac{\mathrm{d}\ln H_s^{cp}}{\mathrm{d}(1/T)}$ [K] | Reference | Type | Note |
|---|---|---|---|---|---|
| 4-chloro-5-methyl-2-nitrophenol $C_7H_6ClNO_3$ (4-chloro-6-nitro-$m$-cresol) [7147-89-9] JBMGJOKJUYGIJH-UHFFFAOYSA-N | $3.6\times10^{-1}$ | | Schwarzenbach et al. (1988) | V | 12 |
| 3-amino-2,5-dichlorobenzoic acid $C_7H_5Cl_2NO_2$ [133-90-4] HSSBORCLYSCBJR-UHFFFAOYSA-N | $2.6\times10^5$ 3.6 $4.5\times10^5$ | | Duchowicz et al. (2020) Mackay et al. (2006d) Duchowicz et al. (2020) | V V Q | 186 |
| 2,3,5,6-tetrachloro-4-nitroanisole $C_7H_3Cl_4NO_3$ (TCNA) [2438-88-2] BGPPUXMKKQMWLV-UHFFFAOYSA-N | $5.2\times10^{-1}$ | | HSDB (2015) | Q | 99 |
| 2,6-dichlorobenzamide $C_7H_5Cl_2NO$ [2008-58-4] JHSPCUHPSIUQRB-UHFFFAOYSA-N | $8.2\times10^3$ | | HSDB (2015) | Q | 99 |
| swep $C_8H_7Cl_2NO$ [1918-18-9] WOZQBERUBLYCEG-UHFFFAOYSA-N | $8.2\times10^2$ | | HSDB (2015) | Q | 99 |
| N-(4-chlorophenyl)acetamide $C_8H_8ClNO$ ($p$-chloroacetanilide) [539-03-7] GGUOCFNAWIODMF-UHFFFAOYSA-N | 2.1 | | HSDB (2015) | Q | 99 |
| methyl 5-chloro-2-nitrobenzoate $C_8H_6ClNO_4$ [51282-49-6] JGBJHRKCUKTQOE-UHFFFAOYSA-N | $9.7\times10^1$ $1.2\times10^2$ $3.7\times10^3$ $6.7\times10^1$ | | Zhang et al. (2010) Zhang et al. (2010) Zhang et al. (2010) Zhang et al. (2010) | Q Q Q Q | 287, 288 287, 289 287, 290 287, 291 |
| 4-chloro-2,5-dimethoxynitrobenzene $C_8H_8ClNO_4$ [6940-53-0] ORLPGMKKCAEWOW-UHFFFAOYSA-N | $1.8\times10^2$ $1.6\times10^1$ $2.0\times10^2$ $3.9\times10^1$ | | Zhang et al. (2010) Zhang et al. (2010) Zhang et al. (2010) Zhang et al. (2010) | Q Q Q Q | 287, 288 287, 289 287, 290 287, 291 |
| chloraniformethan $C_9H_7Cl_5N_2O$ [20856-57-9] REEFSLKDEDEWAO-UHFFFAOYSA-N | $>2.3\times10^{10}$ | | MacBean (2012a) | ? | |



Table A6.8: Chlorocarbons with nitrogen (C, H, O, N, Cl) (...continued)

| Substance<br>Formula<br>(Trivial Name)<br>[CAS Registry Number]<br>InChIKey | $H_s^{cp}$<br>(at $T^{\ominus}$)<br><br>$\left[\dfrac{\text{mol}}{\text{m}^3\,\text{Pa}}\right]$ | $\dfrac{\text{d}\ln H_s^{cp}}{\text{d}(1/T)}$<br><br>[K] | Reference | Type | Note |
|---|---|---|---|---|---|
| monuron<br>$C_9H_{11}ClN_2O$<br>[150-68-5]<br>BMLIZLVNXIYGCK-UHFFFAOYSA-N | $1.7\times10^4$<br>$1.5\times10^4$<br>$3.3\times10^2$<br>$1.7\times10^4$<br>$1.7\times10^4$<br>$1.7\times10^4$ | | HSDB (2015)<br>Mackay et al. (2006d)<br>Suntio et al. (1988)<br>Burkhard and Guth (1981)<br>Abraham et al. (2019)<br>MacBean (2012a) | V<br>V<br>V<br>V<br>Q<br>? | <br><br>12 |
| monolinuron<br>$C_9H_{11}ClN_2O_2$<br>[1746-81-2]<br>LKJPSUCKSLORMF-UHFFFAOYSA-N | $2.1\times10^2$<br>$1.7\times10^2$ | | HSDB (2015)<br>Mackay et al. (2006d) | V<br>V | |
| diuron<br>$C_9H_{10}Cl_2N_2O$<br>[330-54-1]<br>XMTQQYYKAHVGBJ-UHFFFAOYSA-N | $3.5\times10^1$<br>$2.0\times10^4$<br><br>$8.3\times10^2$<br>8.2<br>$4.0\times10^4$<br>$5.2\times10^1$ | | Chao et al. (2017)<br>HSDB (2015)<br>Mackay et al. (2006d)<br>Suntio et al. (1988)<br>Barcelo and Hennion (1997)<br>Abraham et al. (2019)<br>Goodarzi et al. (2010) | M<br>V<br>V<br>V<br>X<br>Q<br>Q | <br><br>558<br>12<br>567<br><br>568 |
| linuron<br>$C_9H_{10}Cl_2N_2O_2$<br>[330-55-2]<br>XKJMBINCVNINCA-UHFFFAOYSA-N | $1.6\times10^3$<br><br>$1.9\times10^2$<br>$5.0\times10^3$<br>1.8<br>6.5<br>$2.2\times10^1$ | | Duchowicz et al. (2020)<br>Mackay et al. (2006d)<br>Suntio et al. (1988)<br>MacBean (2012b)<br>Barcelo and Hennion (1997)<br>Duchowicz et al. (2020)<br>Goodarzi et al. (2010) | V<br>V<br>V<br>X<br>X<br>Q<br>Q | 186<br>558<br>12<br>350<br>567<br><br>568, 571 |
| propanil<br>$C_9H_9Cl_2NO$<br>[709-98-8]<br>LFULEKSKNZEWOE-UHFFFAOYSA-N | $1.3\times10^1$<br>$5.8\times10^3$<br>$5.8\times10^3$<br>$1.8\times10^2$<br>$2.8\times10^2$<br>2.7<br>2.8<br>$2.2\times10^3$<br>$8.0\times10^2$<br>$3.8\times10^3$<br>$8.4\times10^3$<br>$1.2\times10^{-1}$ | | Chao et al. (2017)<br>Duchowicz et al. (2020)<br>HSDB (2015)<br>Mackay et al. (2006d)<br>Suntio et al. (1988)<br>Barcelo and Hennion (1997)<br>Duchowicz et al. (2020)<br>Zhang et al. (2010)<br>Zhang et al. (2010)<br>Zhang et al. (2010)<br>Zhang et al. (2010)<br>Goodarzi et al. (2010) | M<br>V<br>V<br>V<br>V<br>X<br>Q<br>Q<br>Q<br>Q<br>Q<br>Q | <br>186<br><br><br>12<br>567<br><br>287, 288<br>287, 289<br>287, 290<br>287, 291<br>568, 571 |
| methazole<br>$C_9H_6Cl_2N_2O_3$<br>[20354-26-1]<br>LRUUNMYPIBZBQH-UHFFFAOYSA-N | $4.3\times10^1$<br>$4.8\times10^4$<br>$4.3\times10^1$ | | HSDB (2015)<br>Hilal et al. (2008)<br>MacBean (2012a) | V<br>Q<br>? | |



Table A6.8: Chlorocarbons with nitrogen (C, H, O, N, Cl) (. . . continued)

| Substance Formula (Trivial Name) [CAS Registry Number] InChIKey | $H_s^{cp}$ (at $T^\ominus$) $\left[\dfrac{\text{mol}}{\text{m}^3\,\text{Pa}}\right]$ | $\dfrac{\text{d}\ln H_s^{cp}}{\text{d}(1/T)}$ [K] | Reference | Type | Note |
|---|---|---|---|---|---|
| 1,4-naphthalenedione, 2-amino-3-chloro- $C_{10}H_6ClNO_2$ (quinoclamine) [2797-51-5] OBLNWSCLAYSJJR-UHFFFAOYSA-N | $3.3\times10^4$ | | Maniere et al. (2011) | ? | 241, 165 |
| chlortoluron $C_{10}H_{13}ClN_2O$ (chlorotoluron) [15545-48-9] JXCGFZXSOMJFOA-UHFFFAOYSA-N | $7.0\times10^4$ $1.9\times10^4$ $1.9\times10^2$ $3.4\times10^1$ | | HSDB (2015) Mackay et al. (2006d) Barcelo and Hennion (1997) Goodarzi et al. (2010) | V V X Q | 567 568, 571 |
| metoxuron $C_{10}H_{13}ClN_2O_2$ [19937-59-8] DSRNRYQBBJQVCW-UHFFFAOYSA-N | $6.9\times10^2$ $6.9\times10^2$ 6.6 $2.0\times10^2$ $1.2\times10^1$ | | Duchowicz et al. (2020) Mackay et al. (2006d) Barcelo and Hennion (1997) Duchowicz et al. (2020) Goodarzi et al. (2010) | V V X Q Q | 186 567 568 |
| chlorpropham $C_{10}H_{12}ClNO_2$ [101-21-3] CWJSHJJYOPWUGX-UHFFFAOYSA-N | $2.3\times10^1$ $1.7\times10^1$ $1.7\times10^1$ $4.8\times10^2$ $4.7\times10^1$ $5.4\times10^{-2}$ 2.9 | | Watanabe (1993) Duchowicz et al. (2020) HSDB (2015) Mackay et al. (2006d) Suntio et al. (1988) Barcelo and Hennion (1997) Duchowicz et al. (2020) Goodarzi et al. (2010) | M V V V V X Q Q | 186 558 12 567 568, 571 |
| pyrazon $C_{10}H_8ClN_3O$ [1698-60-8] WYKYKTKDBLFHCY-UHFFFAOYSA-N | $3.0\times10^4$ $3.0\times10^4$ $2.3\times10^{-1}$ $2.3\times10^{-1}$ $8.3\times10^4$ $1.9\times10^9$ | | Duchowicz et al. (2020) HSDB (2015) Mackay et al. (2006d) Suntio et al. (1988) Duchowicz et al. (2020) Maniere et al. (2011) | V V V V Q ? | 186 12 12, 165 |
| 2,4-D dimethylamine $C_{10}H_{13}Cl_2NO_3$ ((2,4-dichlorophenoxy)acetic acid dimethylamine) [2008-39-1] IUQJDHJVPLLKFL-UHFFFAOYSA-N | $2.1\times10^{10}$ $4.9\times10^1$ $7.0\times10^{10}$ | | Duchowicz et al. (2020) Duchowicz et al. (2020) HSDB (2015) | V Q Q | 186 99 |
| 3',4'-dichlorocyclopropanecarboxanilide $C_{10}H_9Cl_2NO$ (cypromid) [2759-71-9] PLQDLOBGKJCDSZ-UHFFFAOYSA-N | $3.8\times10^3$ | | HSDB (2015) | Q | 99 |



Table A6.8: Chlorocarbons with nitrogen (C, H, O, N, Cl) (...continued)

| Substance<br>Formula<br>(Trivial Name)<br>[CAS Registry Number]<br>InChIKey | $H_s^{cp}$<br>(at $T^{\ominus}$)<br>$\left[\dfrac{\text{mol}}{\text{m}^3\,\text{Pa}}\right]$ | $\dfrac{\mathrm{d}\ln H_s^{cp}}{\mathrm{d}(1/T)}$<br><br>[K] | Reference | Type | Note |
|---|---|---|---|---|---|
| triazoxide<br>$C_{10}H_6N_5OCl$<br>[72459-58-6]<br>IQGKIPDJXCAMSM-UHFFFAOYSA-N | $4.6\times10^5$<br>$7.9\times10^5$ | | Duchowicz et al. (2020)<br>Duchowicz et al. (2020) | V<br>Q | 186 |
| chlorbufam<br>$C_{11}H_{10}ClNO_2$<br>[1967-16-4]<br>ULBXWWGWDPVHAO-UHFFFAOYSA-N | $1.1\times10^3$<br>$1.1\times10^3$ | | HSDB (2015)<br>MacBean (2012a) | V<br>? | |
| zarilamid<br>$C_{11}H_{11}N_2O_2Cl$<br>[84527-51-5]<br>VLBZAQJMGULJIU-UHFFFAOYSA-N | $1.5\times10^5$ | | MacBean (2012a) | ? | |
| chloramphenicol<br>$C_{11}H_{12}Cl_2N_2O_5$<br>[56-75-7]<br>WIIZWVCIJKGZOK-IUCAKERBSA-N | $4.3\times10^{12}$ | | HSDB (2015) | Q | 99 |
| cloethocarb<br>$C_{11}H_{14}ClNO_4$<br>[51487-69-5]<br>PITWUHDDNUVBPT-UHFFFAOYSA-N | $5.0\times10^5$ | | MacBean (2012a) | ? | |
| formetanate hydrochloride<br>$C_{11}H_{16}ClN_3O_2$<br>[23422-53-9]<br>MYPKGPZHHQEODQ-UHFFFAOYSA-N | $4.3\times10^{13}$<br>$2.0\times10^9$ | | HSDB (2015)<br>Maniere et al. (2011) | Q<br>? | 99<br>241, 165 |
| cyclanilide<br>$C_{11}H_9Cl_2NO_3$<br>[113136-77-9]<br>GLWWLNJJJCTFMZ-UHFFFAOYSA-N | $1.4\times10^4$<br>$1.2\times10^4$<br>$1.2\times10^5$<br>$1.4\times10^4$ | | MacBean (2012b)<br>Keshavarz et al. (2022)<br>Duchowicz et al. (2020)<br>Duchowicz et al. (2020) | X<br>Q<br>Q<br>? | 350<br><br><br>185, 21 |
| propachlor<br>$C_{11}H_{14}ClNO$<br>[1918-16-7]<br>MFOUDYKPLGXPGO-UHFFFAOYSA-N | $2.7\times10^1$<br>$9.1\times10^1$<br>$9.1\times10^1$<br>$9.0\times10^{-1}$<br>$3.7\times10^{-1}$ | | HSDB (2015)<br>Mackay et al. (2006d)<br>Suntio et al. (1988)<br>Barcelo and Hennion (1997)<br>Goodarzi et al. (2010) | V<br>V<br>V<br>X<br>Q | <br><br>12<br>567<br>568 |
| barban<br>$C_{11}H_9Cl_2NO_2$<br>[101-27-9]<br>MCOQHIWZJUDQIC-UHFFFAOYSA-N | $8.4\times10^2$<br>$8.2\times10^2$<br>$8.5\times10^2$<br>$3.3\times10^{-1}$<br>$8.5\times10^2$ | | Duchowicz et al. (2020)<br>HSDB (2015)<br>Mackay et al. (2006d)<br>Duchowicz et al. (2020)<br>MacBean (2012a) | V<br>V<br>V<br>Q<br>? | 186 |
| aclonifen<br>$C_{12}H_9ClN_2O_3$<br>[74070-46-5]<br>DDBMQDADIHOWIC-UHFFFAOYSA-N | $5.9\times10^2$<br>$7.1\times10^3$<br>$3.3\times10^2$ | | Duchowicz et al. (2020)<br>Duchowicz et al. (2020)<br>Maniere et al. (2011) | V<br>Q<br>? | 186<br><br>12, 165 |



Table A6.8: Chlorocarbons with nitrogen (C, H, O, N, Cl) (...continued)

| Substance Formula (Trivial Name) [CAS Registry Number] InChIKey | $H_s^{cp}$ (at $T^{\ominus}$) $\left[\dfrac{\text{mol}}{\text{m}^3\,\text{Pa}}\right]$ | $\dfrac{\text{d}\ln H_s^{cp}}{\text{d}(1/T)}$ [K] | Reference | Type | Note |
|---|---|---|---|---|---|
| propyzamide $C_{12}H_{11}Cl_2NO$ (pronamide) [23950-58-5] PHNUZKMIPFFYSO-UHFFFAOYSA-N | $1.0\times10^3$ $5.2$ $5.2\times10^{-2}$ $1.5\times10^2$ $6.6\times10^{-2}$ $1.3\times10^8$ | | Duchowicz et al. (2020) HSDB (2015) Barcelo and Hennion (1997) Duchowicz et al. (2020) Goodarzi et al. (2010) Maniere et al. (2011) Mackay et al. (2006d) | V V X Q Q ? W | 186 567 568 12, 165 736 |
| 2,4,6-trichlorophenyl 4-nitrophenyl ether $C_{12}H_6Cl_3NO_3$ (chlornitrofen) [1836-77-7] XQNAUQUKWRBODG-UHFFFAOYSA-N | $>8.1$ | | Kawamoto and Urano (1989) | M | |
| nitrofen $C_{12}H_7Cl_2NO_3$ [1836-75-5] XITQUSLLOSKDTB-UHFFFAOYSA-N | $3.3$ $3.9\times10^1$ $2.8\times10^1$ $1.2\times10^2$ $1.1\times10^2$ | | HSDB (2015) Zhang et al. (2010) Zhang et al. (2010) Zhang et al. (2010) Zhang et al. (2010) | V Q Q Q Q | 287, 288 287, 289 287, 290 287, 291 |
| buturon $C_{12}H_{13}ClN_2O$ [3766-60-7] BYYMILHAKOURNM-UHFFFAOYSA-N | $1.3\times10^4$ | | MacBean (2012a) | ? | |
| triclocarban $C_{13}H_9Cl_3N_2O$ [101-20-2] ICUTUKXCWQYESQ-UHFFFAOYSA-N | $2.2\times10^5$ $2.2\times10^5$ $5.0\times10^3$ $7.2\times10^7$ $1.8\times10^7$ | | HSDB (2015) Zhang et al. (2010) Zhang et al. (2010) Zhang et al. (2010) Zhang et al. (2010) | Q Q Q Q Q | 99 287, 288 287, 289 287, 290 287, 291 |
| dimethachlor $C_{13}H_{18}ClNO_2$ [50563-36-5] SCCDDNKJYDZXMM-UHFFFAOYSA-N | $5.9\times10^3$ | | Maniere et al. (2011) | ? | 165 |
| 3,5-dichloro-N-(3,4-dichlorophenyl)-2-hydroxybenzamide $C_{13}H_7Cl_4NO_2$ [1154-59-2] SJQBHPJLLIJASD-UHFFFAOYSA-N | $2.1\times10^5$ $2.1\times10^5$ $2.3\times10^5$ $3.9\times10^6$ $1.6\times10^5$ | | HSDB (2015) Zhang et al. (2010) Zhang et al. (2010) Zhang et al. (2010) Zhang et al. (2010) | Q Q Q Q Q | 99 287, 288 287, 289 287, 290 287, 291 |
| procymidone $C_{13}H_{11}Cl_2NO_2$ [32809-16-8] QXJKBPAVAHBARF-UHFFFAOYSA-N | $8.5\times10^{-1}$ $8.5\times10^{-1}$ $9.0\times10^{-3}$ $9.7\times10^{-1}$ $1.4\times10^{-2}$ | | Duchowicz et al. (2020) Mackay et al. (2006d) Barcelo and Hennion (1997) Duchowicz et al. (2020) Goodarzi et al. (2010) | V V X Q Q | 186 567 568 |





Table A6.8: Chlorocarbons with nitrogen (C, H, O, N, Cl) (... continued)

| Substance Formula (Trivial Name) [CAS Registry Number] InChIKey | $H_s^{cp}$ (at $T^{\ominus}$) $\left[\dfrac{\text{mol}}{\text{m}^3\,\text{Pa}}\right]$ | $\dfrac{\text{d}\ln H_s^{cp}}{\text{d}(1/T)}$ [K] | Reference | Type | Note |
|---|---|---|---|---|---|
| melphalan $C_{13}H_{18}Cl_2N_2O_2$ [148-82-3] SGDBTWWWUNNDEQ-LBPRGKRZSA-N | $2.3\times10^7$ | | HSDB (2015) | Q | 99 |
| niclosamide $C_{13}H_8Cl_2N_2O_4$ [50-65-7] RJMUSRYZPJIFPJ-UHFFFAOYSA-N | $1.5\times10^4$ | | HSDB (2015) | V | |
| iprodione $C_{13}H_{13}Cl_2N_3O_3$ [36734-19-7] ONUFESLQCSAYKA-UHFFFAOYSA-N | $7.1\times10^2$ $4.0\times10^4$ $9.5\times10^3$ $2.0\times10^1$ $3.2\times10^3$ | | Barcelo and Hennion (1997) Keshavarz et al. (2022) Duchowicz et al. (2020) Goodarzi et al. (2010) Duchowicz et al. (2020) | X Q Q Q ? | 567 568, 569 185, 21 |
| zoxamide $C_{14}H_{16}Cl_3NO_2$ [156052-68-5] SOUGWDPPRBKJEX-UHFFFAOYSA-N | $4.9\times10^3$ $>1.5\times10^2$ | | HSDB (2015) Maniere et al. (2011) | Q ? | 99 241, 165 |
| fenhexamid $C_{14}H_{17}Cl_2NO_2$ [126833-17-8] VDLGAVXLJYLFDH-UHFFFAOYSA-N | $2.0\times10^5$ $2.0\times10^5$ $3.3\times10^6$ $1.1\times10^5$ | | MacBean (2012b) Maniere et al. (2011) Maniere et al. (2011) Maniere et al. (2011) | X ? ? ? | 350 12, 493, 165 12, 573, 165 12, 570, 165 |
| chlorambucil $C_{14}H_{19}Cl_2NO_2$ [305-03-3] JCKYGMPEJWAADB-UHFFFAOYSA-N | $3.7\times10^4$ | | HSDB (2015) | Q | 99 |
| 2-chloro-N-(ethoxymethyl)-N-(2-ethyl-6-methylphenyl)acetamide $C_{14}H_{20}ClNO_2$ (acetochlor) [34256-82-1] VTNQPKFIQCLBDU-UHFFFAOYSA-N | $3.7\times10^4$ | | HSDB (2015) | V | |
| alachlor $C_{14}H_{20}ClNO_2$ [15972-60-8] XCSGPAVHZFQHGE-UHFFFAOYSA-N | $6.7\times10^2$ $9.9\times10^2$ $7.4$ $1.4\times10^2$ $9.0\times10^2$ $1.2\times10^3$ $4.5\times10^2$ $1.6\times10^2$ $3.1\times10^2$ $1.6$ | 9200 | Muir et al. (2004) Muir et al. (2004) Chao et al. (2017) Gautier et al. (2003) Fendinger et al. (1989) Fendinger and Glotfelty (1988) Mackay et al. (2006d) Suntio et al. (1988) Glotfelty et al. (1987) Barcelo and Hennion (1997) | L L M M M M V V V X | 367 366 72 72 12 567 |



Table A6.8: Chlorocarbons with nitrogen (C, H, O, N, Cl) (... continued)

| Substance Formula (Trivial Name) [CAS Registry Number] InChIKey | $H_s^{cp}$ (at $T^\ominus$) $\left[\dfrac{\mathrm{mol}}{\mathrm{m^3\,Pa}}\right]$ | $\dfrac{\mathrm{d}\ln H_s^{cp}}{\mathrm{d}(1/T)}$ [K] | Reference | Type | Note |
|---|---|---|---|---|---|
| | $3.6\times10^{-2}$ | | Goodarzi et al. (2010) | Q | 568, 569 |
| | $3.1\times10^{3}$ | | Hilal et al. (2008) | Q | |
| | $4.5\times10^{1}$ | | Modarresi et al. (2007) | Q | 67 |
| | | 11000 | Kühne et al. (2005) | Q | |
| | $8.2\times10^{4}$ | | Meylan and Howard (1991) | Q | |
| | | 9300 | Kühne et al. (2005) | ? | |
| | $3.1\times10^{2}$ | | Chesters et al. (1989) | ? | |
| bifenox $C_{14}H_9Cl_2NO_3$ [42576-02-3] SUSRORUBZHMPCO-UHFFFAOYSA-N | $9.1\times10^{1}$ | | Duchowicz et al. (2020) | V | 186 |
| | 3.7 | | HSDB (2015) | V | |
| | 3.2 | | Mackay et al. (2006d) | V | |
| | $3.3\times10^{-2}$ | | Barcelo and Hennion (1997) | X | 567 |
| | $6.2\times10^{3}$ | | Duchowicz et al. (2020) | Q | |
| | $3.5\times10^{-2}$ | | Goodarzi et al. (2010) | Q | 568, 571 |
| | $<6.2\times10^{3}$ | | Maniere et al. (2011) | ? | 12, 165 |
| $S$-metolachlor $C_{15}H_{22}ClNO_2$ [87392-12-9] WVQBLGZPHOPPFO-LBPRGKRZSA-N | $4.5\times10^{2}$ | | Maniere et al. (2011) | ? | 165 |
| metolachlor $C_{15}H_{22}ClNO_2$ [51218-45-2] WVQBLGZPHOPPFO-UHFFFAOYSA-N | $7.5\times10^{2}$ | | Muir et al. (2004) | L | 367 |
| | $7.2\times10^{2}$ | | Muir et al. (2004) | L | 366 |
| | $6.2\times10^{2}$ | 15000 | Fogg and Sangster (2003) | L | |
| | $2.1\times10^{2}$ | 10000 | Feigenbrugel et al. (2004a) | M | |
| | $1.3\times10^{2}$ | | Rice et al. (1997b) | M | 12 |
| | $5.7\times10^{2}$ | 15000 | Lau et al. (1995) | M | 737 |
| | $4.3\times10^{2}$ | | Mackay et al. (2006d) | V | |
| | $4.1\times10^{2}$ | | Otto et al. (1997) | V | |
| | $1.1\times10^{3}$ | | Glotfelty et al. (1987) | V | |
| | $1.1\times10^{3}$ | | Burkhard and Guth (1981) | V | |
| | $1.1\times10^{1}$ | | Barcelo and Hennion (1997) | X | 567 |
| | $1.2\times10^{3}$ | | Rice et al. (1997b) | C | |
| | $7.9\times10^{-2}$ | | Goodarzi et al. (2010) | Q | 568 |
| | $6.2\times10^{3}$ | | Hilal et al. (2008) | Q | |
| | $1.7\times10^{2}$ | | Modarresi et al. (2007) | Q | 67 |
| | | 12000 | Kühne et al. (2005) | Q | |
| | | 10000 | Kühne et al. (2005) | ? | |
| | $1.1\times10^{3}$ | | Chesters et al. (1989) | ? | 12 |
| clonitralid $C_{15}H_{15}Cl_2N_3O_5$ [1420-04-8] XYCDHXSQODHSLG-UHFFFAOYSA-N | $>2.6\times10^{4}$ | | HSDB (2015) | V | |
| chloroxuron $C_{15}H_{15}ClN_2O_2$ [1982-47-4] IVUXTESCPZUGJC-UHFFFAOYSA-N | $2.4\times10^{4}$ | | HSDB (2015) | V | |
| | $5.3\times10^{4}$ | | MacBean (2012a) | ? | |



Table A6.8: Chlorocarbons with nitrogen (C, H, O, N, Cl) (... continued)

| Substance Formula (Trivial Name) [CAS Registry Number] InChIKey | $H_s^{cp}$ (at $T^\ominus$) $\left[\dfrac{\mathrm{mol}}{\mathrm{m}^3\,\mathrm{Pa}}\right]$ | $\dfrac{\mathrm{d}\ln H_s^{cp}}{\mathrm{d}(1/T)}$ [K] | Reference | Type | Note |
|---|---|---|---|---|---|
| CGA 80000 C$_{15}$H$_{18}$ClNO$_4$ [67932-85-8] FHZMAAGOSXDIBJ-UHFFFAOYSA-N | $4.4\times10^6$ | | MacBean (2012a) | ? | |
| oxadiargyl C$_{15}$H$_{14}$N$_2$O$_3$Cl$_2$ [39807-15-3] DVOODWOZJVJKQR-UHFFFAOYSA-N | $4.3\times10^2$ $8.2\times10^1$ | | Duchowicz et al. (2020) Duchowicz et al. (2020) | V Q | 186 |
| diniconazole C$_{15}$H$_{17}$Cl$_2$N$_3$O [83657-24-3] FBOUIAKEJMZPQG-UHFFFAOYSA-N | $2.5\times10^1$ $2.8\times10^2$ | | Duchowicz et al. (2020) Duchowicz et al. (2020) | V Q | 186 |
| pethoxamid C$_{16}$H$_{22}$ClNO$_2$ [106700-29-2] CSWIKHNSBZVWNQ-UHFFFAOYSA-N | $8.5\times10^2$ | | Maniere et al. (2011) | ? | 12, 165 |
| pigment red 4 C$_{16}$H$_{10}$ClN$_3$O$_3$ [2814-77-9] XLTMWFMRJZDFFD-VHEBQXMUSA-N | $1.1\times10^7$ | | HSDB (2015) | Q | 99 |
| darendoside b C$_{17}$H$_{15}$Cl$_2$N$_5$O$_2$ [13301-61-6] KHZRTXVUEZJYNE-UHFFFAOYSA-N | $2.7\times10^7$ $5.0\times10^6$ $2.5\times10^7$ $7.3\times10^4$ | | Zhang et al. (2010) Zhang et al. (2010) Zhang et al. (2010) Zhang et al. (2010) | Q Q Q Q | 287, 288 287, 289 287, 290 287, 291 |
| butenachlor C$_{17}$H$_{24}$NO$_2$Cl [87310-56-3] HZDIJTXDRLNTIS-DAXSKMNVSA-N | $1.0\times10^2$ | | MacBean (2012a) | ? | |
| butachlor C$_{17}$H$_{26}$ClNO$_2$ [23184-66-9] HKPHPIREJKHECO-UHFFFAOYSA-N | $1.6\times10^2$ $1.2\times10^2$ $6.9\times10^2$ $1.7\times10^2$ | | Watanabe (1993) Mackay et al. (2006d) Hilal et al. (2008) Modarresi et al. (2007) | M V Q Q | 67 |
| pretilachlor C$_{17}$H$_{26}$ClNO$_2$ [51218-49-6] YLPGTOIOYRQOHV-UHFFFAOYSA-N | $4.5\times10^3$ $1.8\times10^2$ | | Hilal et al. (2008) Modarresi et al. (2007) | Q Q | 67 |
| halofenozide C$_{18}$H$_{19}$ClN$_2$O$_2$ [112226-61-6] CNKHSLKYRMDDNQ-UHFFFAOYSA-N | $2.7\times10^5$ | | HSDB (2015) | Q | 99 |





Table A6.8: Chlorocarbons with nitrogen (C, H, O, N, Cl) (...continued)

| Substance Formula (Trivial Name) [CAS Registry Number] InChIKey | $H_s^{cp}$ (at $T^\ominus$) $\left[\dfrac{\mathrm{mol}}{\mathrm{m^3\,Pa}}\right]$ | $\dfrac{\mathrm{d}\ln H_s^{cp}}{\mathrm{d}(1/T)}$ [K] | Reference | Type | Note |
|---|---|---|---|---|---|
| benzoximate $C_{18}H_{18}ClNO_5$ [29104-30-1] BZMIHNKNQJJVRO-UHFFFAOYSA-N | $1.8\times10^2$ $1.5\times10^5$ | | Duchowicz et al. (2020) Duchowicz et al. (2020) | V Q | 186 |
| pencycuron $C_{19}H_{21}ClN_2O$ [66063-05-6] OGYFATSSENRIKG-UHFFFAOYSA-N | $2.0\times10^6$ | | Maniere et al. (2011) | ? | 12, 165 |
| valifenalate $C_{19}H_{27}ClN_2O_5$ [283159-90-0] DBXFMOWZRXXBRN-LWKPJOBUSA-N | $6.2\times10^5$ | | Maniere et al. (2011) | ? | 12, 738, 165 |
| pyrazoxyfen $C_{20}H_{16}Cl_2N_2O_3$ [71561-11-0] FKERUJTUOYLBKB-UHFFFAOYSA-N | $4.7\times10^4$ $1.9\times10^4$ | | Duchowicz et al. (2020) Duchowicz et al. (2020) | V Q | 186 |
| $\alpha$-cypermethrin $C_{22}H_{19}Cl_2NO_3$ [67375-30-8] KAATUXNTWXVJKI-DXCJPMOASA-N | 1.0 $1.0\times10^2$ $1.9\times10^1$ | | HSDB (2015) Mackay et al. (2006d) Maniere et al. (2011) | V V ? | 12, 165 |
| cypermethrin-$\beta$ $C_{22}H_{19}Cl_2NO_3$ [1224510-29-5] KAATUXNTWXVJKI-FLXSOZOKSA-N | $5.6\times10^1$ | | Ebert et al. (2023) | ? | 318 |
| $\beta$-cypermethrin $C_{22}H_{19}Cl_2NO_3$ [65731-84-2] KAATUXNTWXVJKI-NSHGMRRFSA-N | | | Mackay et al. (2006d) | V | 558 |
| $\delta$-cypermethrin $C_{22}H_{19}Cl_2NO_3$ (cypermethrin; alphamethrin) [52315-07-8] KAATUXNTWXVJKI-UHFFFAOYSA-N | $4.1\times10^1$ $4.3\times10^2$ $1.2\times10^1$ $2.5\times10^1$ $1.0\times10^1$ $4.2\times10^1$ $5.0\times10^7$ | | HSDB (2015) Mackay et al. (2006d) Siebers and Mattusch (1996) Barcelo and Hennion (1997) Goodarzi et al. (2010) Maniere et al. (2011) Maniere et al. (2011) | V V V X Q ? ? | 12 567 568, 569 12, 165 241, 165 |
| $\theta$-cypermethrin $C_{22}H_{19}Cl_2NO_3$ [71697-59-1] KAATUXNTWXVJKI-GGPKGHCWSA-N | $6.1\times10^2$ | | Ebert et al. (2023) | ? | 318 |



Table A6.8: Chlorocarbons with nitrogen (C, H, O, N, Cl) (... continued)

| Substance<br>Formula<br>(Trivial Name)<br>[CAS Registry Number]<br>InChIKey | $H_s^{cp}$<br>(at $T^{\ominus}$)<br>$\left[\dfrac{\mathrm{mol}}{\mathrm{m^3\,Pa}}\right]$ | $\dfrac{\mathrm{d}\ln H_s^{cp}}{\mathrm{d}(1/T)}$<br><br>[K] | Reference | Type | Note |
|---|---|---|---|---|---|
| $\zeta$-cypermethrin<br>$C_{22}H_{19}Cl_2NO_3$<br>[1315501-18-8]<br>KAATUXNTWXVJKI-QPIRBTGLSA-N | $4.3\times10^2$ | | Ebert et al. (2023) | ? | 365 |
| ochratoxin C<br>$C_{22}H_{22}ClNO_6$<br>[4865-85-4]<br>BPZZWRPHVVDAPT-PXAZEXFGSA-N | $7.6\times10^8$ | | HSDB (2015) | Q | 99 |
| quizalofop-p-tefuryl<br>$C_{22}H_{21}N_2O_5Cl$<br>[119738-06-6]<br>BBKDWPHJZANJGB-UHFFFAOYSA-N | $1.8\times10^4$<br>$3.4\times10^6$ | | Duchowicz et al. (2020)<br>Duchowicz et al. (2020) | V<br>Q | 186 |
| mandipropamid<br>$C_{23}H_{22}ClNO_4$<br>[374726-62-2]<br>KWLVWJPJKJMCSH-UHFFFAOYSA-N | $1.1\times10^4$<br>$>1.1\times10^4$ | | HSDB (2015)<br>Maniere et al. (2011) | V<br>? | 165 |
| fenvalerate<br>$C_{25}H_{22}ClNO_3$<br>[51630-58-1]<br>NYPJDWWKZLNGGM-UHFFFAOYSA-N | $2.9\times10^2$<br>$2.9\times10^2$<br>$4.7\times10^1$<br>$7.0\times10^1$<br>$3.2\times10^3$ | | Duchowicz et al. (2020)<br>HSDB (2015)<br>Mackay et al. (2006d)<br>Cotham and Bidleman (1989)<br>Duchowicz et al. (2020) | V<br>V<br>V<br>V<br>Q | 186 |
| esfenvalerate<br>$C_{25}H_{22}ClNO_3$<br>[66230-04-4]<br>NYPJDWWKZLNGGM-ZEQRLZLVSA-N | $2.4\times10^1$<br>$2.4\times10^1$<br>$3.2\times10^3$<br>$2.0\times10^3$ | | Duchowicz et al. (2020)<br>HSDB (2015)<br>Duchowicz et al. (2020)<br>Maniere et al. (2011) | V<br>V<br>Q<br>? | 186<br><br><br>241, 165 |
| clopyralid<br>$C_6H_3Cl_2NO_2$<br>[1702-17-6]<br>HUBANNPOLNYSAD-UHFFFAOYSA-N | $3.3\times10^3$<br>$3.2\times10^2$<br>$1.8\times10^3$<br>$3.6\times10^1$<br>$3.0\times10^9$<br>$4.5\times10^{10}$<br><br>$6.2\times10^{10}$<br><br>$5.6\times10^{10}$ | | Duchowicz et al. (2020)<br>Barcelo and Hennion (1997)<br>Duchowicz et al. (2020)<br>Goodarzi et al. (2010)<br>Maniere et al. (2011)<br>Maniere et al. (2011)<br><br>Maniere et al. (2011)<br><br>Maniere et al. (2011) | V<br>X<br>Q<br>Q<br>?<br>?<br><br>?<br><br>? | 186<br>567<br><br>568<br>12, 165<br>12, 570,<br>165<br>12, 573,<br>165<br>12, 493,<br>165 |
| picloram<br>$C_6H_3Cl_3N_2O_2$<br>[1918-02-1]<br>NQQVFXUMIDALNH-UHFFFAOYSA-N | $3.0\times10^4$<br>$2.9\times10^4$<br>$2.9\times10^2$<br>$7.7\times10^6$<br>$2.5$<br>$9.0\times10^4$<br>$1.6\times10^8$<br>$1.1\times10^3$ | | Mackay et al. (2006d)<br>Suntio et al. (1988)<br>Barcelo and Hennion (1997)<br>Zhang et al. (2010)<br>Zhang et al. (2010)<br>Zhang et al. (2010)<br>Zhang et al. (2010)<br>Goodarzi et al. (2010) | V<br>V<br>X<br>Q<br>Q<br>Q<br>Q<br>Q | <br>12<br>567<br>287, 288<br>287, 289<br>287, 290<br>287, 291<br>568, 571 |



Table A6.8: Chlorocarbons with nitrogen (C, H, O, N, Cl) (…continued)

| Substance<br>Formula<br>(Trivial Name)<br>[CAS Registry Number]<br>InChIKey | $H_s^{cp}$<br>(at $T^{\ominus}$)<br><br>$\left[\dfrac{\mathrm{mol}}{\mathrm{m^3\,Pa}}\right]$ | $\dfrac{\mathrm{d}\ln H_s^{cp}}{\mathrm{d}(1/T)}$<br><br>[K] | Reference | Type | Note |
|---|---|---|---|---|---|
| | $2.1\times10^4$ | | Maniere et al. (2011) | ? | 241, 165 |
| aminopyralid<br>$C_6H_4Cl_2N_2O_2$<br><br>[150114-71-9]<br>NIXXQNOQHKNPEJ-UHFFFAOYSA-N | $5.8\times10^6$<br>$1.0\times10^{11}$ | | HSDB (2015)<br>Maniere et al. (2011) | Q<br>? | 99<br>241, 493, 165 |
| 3,4,5,6-tetrachloropyridine-2-<br>carboxylic<br>acid<br>$C_6HCl_4NO_2$<br>[10469-09-7]<br>GXFRQLQUKBSYQL-UHFFFAOYSA-N | $3.7\times10^3$<br><br><br>2.4<br>$1.2\times10^2$<br>$4.1\times10^4$ | | Zhang et al. (2010)<br><br><br>Zhang et al. (2010)<br>Zhang et al. (2010)<br>Zhang et al. (2010) | Q<br><br><br>Q<br>Q<br>Q | 287, 288<br><br><br>287, 289<br>287, 290<br>287, 291 |
| [(3,5,6-trichloro-2-pyridinyl)oxy]-<br>acetic<br>acid<br>$C_7H_4Cl_3NO_3$<br>(triclopyr)<br>[55335-06-3]<br>REEQLXCGVXDJSQ-UHFFFAOYSA-N | $1.0\times10^4$<br><br><br>$1.0\times10^4$<br>$1.2\times10^4$<br>$3.2\times10^3$<br>$1.1\times10^4$<br>$2.1\times10^5$<br><br>$2.3\times10^5$ | | Duchowicz et al. (2020)<br><br><br>HSDB (2015)<br>Armbrust (2000)<br>Duchowicz et al. (2020)<br>Maniere et al. (2011)<br>Maniere et al. (2011)<br><br>Maniere et al. (2011) | V<br><br><br>V<br>C<br>Q<br>?<br>?<br><br>? | 186<br><br><br><br><br><br>12, 165<br>12, 570, 165<br><br>12, 493, 165 |
| clopidol<br>$C_7H_7Cl_2NO$<br>[2971-90-6]<br>ZDPIZLCVJAAHHR-UHFFFAOYSA-N | $9.9\times10^3$ | | HSDB (2015) | Q | 99 |
| [(3,5,6-trichloro-2-pyridinyl)oxy]-<br>acetic acid, methyl<br>ester<br>$C_8H_6Cl_3NO_3$<br>[60825-26-5]<br>MNYBZEHWPRTNJY-UHFFFAOYSA-N | 6.0<br><br><br>$3.1\times10^1$<br>$4.6\times10^3$<br>$3.5\times10^2$ | | Zhang et al. (2010)<br><br><br>Zhang et al. (2010)<br>Zhang et al. (2010)<br>Zhang et al. (2010) | Q<br><br><br>Q<br>Q<br>Q | 287, 288<br><br><br>287, 289<br>287, 290<br>287, 291 |
| aminocyclopyrachlor<br>$C_8H_8ClN_3O_2$<br>[858956-08-8]<br>KWAIHLIXESXTJL-UHFFFAOYSA-N | $1.3\times10^7$ | | Ebert et al. (2023) | ? | 318 |
| uracil mustard<br>$C_8H_{11}Cl_2N_3O_2$<br>[66-75-1]<br>IDPUKCWIGUEADI-UHFFFAOYSA-N | $2.5\times10^7$ | | HSDB (2015) | Q | 99 |



Table A6.8: Chlorocarbons with nitrogen (C, H, O, N, Cl) (. . . continued)

| Substance Formula (Trivial Name) [CAS Registry Number] InChIKey | $H_s^{cp}$ (at $T^{\ominus}$) $\left[\dfrac{\mathrm{mol}}{\mathrm{m^3\,Pa}}\right]$ | $\dfrac{\mathrm{d}\ln H_s^{cp}}{\mathrm{d}(1/T)}$ [K] | Reference | Type | Note |
|---|---|---|---|---|---|
| imidacloprid $C_9H_{10}ClN_5O_2$ [138261-41-3] YWTYJOPNNQFBPC-UHFFFAOYSA-N | $4.9 \times 10^9$ $5.9 \times 10^9$ | | Armbrust (2000) Maniere et al. (2011) | C ? | 12, 165 |
| ethyl [(3,5,6-trichloro-2-pyridinyl)oxy]acetate $C_9H_8Cl_3NO_3$ [60825-27-6] KXAVVWXJUDQGDA-UHFFFAOYSA-N | 4.5 $1.7 \times 10^1$ $2.3 \times 10^1$ $3.1 \times 10^2$ | | Zhang et al. (2010) Zhang et al. (2010) Zhang et al. (2010) Zhang et al. (2010) | Q Q Q Q | 287, 288 287, 289 287, 290 287, 291 |
| N-methyl-3,4,5,6-tetrachlorophthalimide $C_9H_3Cl_4NO_2$ [14737-80-5] OHCSZUQRNNNMRG-UHFFFAOYSA-N | $1.5 \times 10^3$ $1.2 \times 10^3$ $4.1 \times 10^1$ $3.1 \times 10^3$ | | Zhang et al. (2010) Zhang et al. (2010) Zhang et al. (2010) Zhang et al. (2010) | Q Q Q Q | 287, 288 287, 289 287, 290 287, 291 |
| terbacil $C_9H_{13}ClN_2O_2$ [5902-51-2] NBQCNZYJJMBDKY-UHFFFAOYSA-N | $5.2 \times 10^4$ $6.5 \times 10^4$ $7.9 \times 10^4$ $5.6 \times 10^4$ $5.5 \times 10^2$ $4.4 \times 10^2$ | | HSDB (2015) Mackay et al. (2006d) Mackay et al. (2006d) Suntio et al. (1988) Barcelo and Hennion (1997) Goodarzi et al. (2010) | V V V V X Q | 12 567 568 |
| eglinazine-ethyl $C_9H_{14}ClN_5O_2$ [6616-80-4] YESXTECNXIKUMM-UHFFFAOYSA-N | $4.1 \times 10^4$ | | Ebert et al. (2023) | ? | 318 |
| proglinazine ethyl ester $C_{10}H_{16}ClN_5O_2$ [68228-18-2] QQADVTSTCZBBOE-UHFFFAOYSA-N | $1.0 \times 10^4$ | | Ebert et al. (2023) | ? | 318 |
| triforine $C_{10}H_{14}Cl_6N_4O_2$ [26644-46-2] RROQIUMZODEXOR-UHFFFAOYSA-N | $2.6 \times 10^3$ $2.6 \times 10^3$ | | HSDB (2015) Mackay et al. (2006d) | V V | |
| drazoxolon $C_{10}H_8ClN_3O_2$ [5707-69-7] OOTHTARUZHONSW-UHFFFAOYSA-N | $1.4 \times 10^2$ | | Ebert et al. (2023) | ? | 318 |
| anagrelide $C_{10}H_7Cl_2N_3O$ [68475-42-3] OTBXOEAOVRKTNQ-UHFFFAOYSA-N | $3.7 \times 10^7$ | | HSDB (2015) | Q | 99 |





Table A6.8: Chlorocarbons with nitrogen (C, H, O, N, Cl) (...continued)

| Substance<br>Formula<br>(Trivial Name)<br>[CAS Registry Number]<br>InChIKey | $H_s^{cp}$<br>(at $T^{\ominus}$)<br>$\left[\dfrac{\text{mol}}{\text{m}^3\,\text{Pa}}\right]$ | $\dfrac{\text{d}\ln H_s^{cp}}{\text{d}(1/T)}$<br><br>[K] | Reference | Type | Note |
|---|---|---|---|---|---|
| fenpiclonil<br>$C_{11}H_6Cl_2N_2$<br>[74738-17-3]<br>FKLFBQCQQYDUAM-UHFFFAOYSA-N | $1.8\times10^3$<br>$5.4\times10^2$<br>$1.9\times10^3$ | | Duchowicz et al. (2020)<br>Duchowicz et al. (2020)<br>MacBean (2012a) | V<br>Q<br>? | 186 |
| benoxacor<br>$C_{11}H_{11}Cl_2NO_2$<br>[98730-04-2]<br>PFJJMJDEVDLPNE-UHFFFAOYSA-N | $1.3\times10^2$<br>$2.9\times10^2$<br>$1.3\times10^2$ | | Duchowicz et al. (2020)<br>Duchowicz et al. (2020)<br>Maniere et al. (2011) | V<br>Q<br>? | 186<br><br>12, 165 |
| ethychlozate<br>$C_{11}H_{11}ClN_2O_2$<br>[27512-72-7]<br>GLPZEHFBLBYFHN-UHFFFAOYSA-N | $1.6\times10^4$ | | Ebert et al. (2023) | ? | 318 |
| quinmerac<br>$C_{11}H_8ClNO_2$<br>[90717-03-6]<br>ALZOLUNSQWINIR-UHFFFAOYSA-N | $1.0\times10^{10}$ | | Maniere et al. (2011) | ? | 241, 165 |
| quinonamid<br>$C_{12}H_6Cl_3NO_3$<br>[27541-88-4]<br>ZIEWAMOXCOLNSJ-UHFFFAOYSA-N | $8.0\times10^2$ | | Ebert et al. (2023) | ? | 365 |
| fenchlorazole-ethyl<br>$C_{12}H_8N_3O_2Cl_5$<br>[103112-35-2]<br>GMBRUAIJEFRHFQ-UHFFFAOYSA-N | $2.7\times10^3$ | | MacBean (2012a) | ? | 12 |
| vinclozoline<br>$C_{12}H_9Cl_2NO_3$<br>(vinclozolin)<br>[50471-44-8]<br>FSCWZHGZWWDELK-UHFFFAOYSA-N | $5.8\times10^2$<br>$2.6\times10^5$<br>$9.1\times10^1$<br>7.6<br>1.7 | | HSDB (2015)<br>Mackay et al. (2006d)<br>Siebers et al. (1994)<br>Barcelo and Hennion (1997)<br>Goodarzi et al. (2010) | V<br>V<br>V<br>X<br>Q | <br><br><br>567<br>568 |
| forchlorfenuron<br>$C_{12}H_{10}ClN_3O$<br>[68157-60-8]<br>GPXLRLUVLMHHIK-UHFFFAOYSA-N | $3.4\times10^6$<br>$3.5\times10^6$<br>$1.7\times10^1$<br>$3.4\times10^6$ | | Duchowicz et al. (2020)<br>MacBean (2012b)<br>Duchowicz et al. (2020)<br>Maniere et al. (2011) | V<br>X<br>Q<br>? | 186<br>350<br><br>241, 165 |
| azaconazole<br>$C_{12}H_{11}Cl_2N_3O_2$<br>[60207-31-0]<br>AKNQMEBLVAMSNZ-UHFFFAOYSA-N | $5.1\times10^4$ | | Ebert et al. (2023) | ? | 318 |
| myclozolin<br>$C_{12}H_{11}NO_4Cl_2$<br>[54864-61-8]<br>FTCOKXNKPOUEFH-UHFFFAOYSA-N | $3.7\times10^2$ | | MacBean (2012a) | ? | |





Table A6.8: Chlorocarbons with nitrogen (C, H, O, N, Cl) (... continued)

| Substance Formula (Trivial Name) [CAS Registry Number] InChIKey | $H_s^{cp}$ (at $T^\ominus$) $\left[\dfrac{\text{mol}}{\text{m}^3\,\text{Pa}}\right]$ | $\dfrac{\text{d}\ln H_s^{cp}}{\text{d}(1/T)}$ [K] | Reference | Type | Note |
|---|---|---|---|---|---|
| triclopyr-butotyl $C_{13}H_{16}Cl_3NO_4$ [64700-56-7] IVDRCZNHVGQBHZ-UHFFFAOYSA-N | $3.4\times10^2$ | | Maniere et al. (2011) | ? | 12, 165 |
| clofencet $C_{13}H_{11}ClN_2O_3$ [129025-54-3] PIZCXVUFSNPNON-UHFFFAOYSA-N | $>1.9\times10^8$ $>2.3\times10^{10}$ | | HSDB (2015) MacBean (2012a) | V ? | |
| chlozolinate $C_{13}H_{11}NO_5Cl_2$ [84332-86-5] IGUYEXXAGBDLLX-UHFFFAOYSA-N | $7.6\times10^3$ $4.7\times10^2$ $2.8\times10^1$ $2.8\times10^1$ $4.4\times10^2$ | | Duchowicz et al. (2020) Duchowicz et al. (2020) Duchowicz et al. (2020) Duchowicz et al. (2020) MacBean (2012a) | V V Q Q ? | 186 186 |
| monalide $C_{13}H_{18}ClNO$ [7287-36-7] KXGYBSNVFXBPNO-UHFFFAOYSA-N | $4.0\times10^2$ | | MacBean (2012a) | ? | |
| chlomethoxyfen $C_{13}H_9Cl_2NO_4$ [32861-85-1] DXXVCXKMSWHGTF-UHFFFAOYSA-N | $5.1\times10^{-1}$ | | Ebert et al. (2023) | ? | 318 |
| metazachlor $C_{14}H_{16}ClN_3O$ [67129-08-2] STEPQTYSZVCJPV-UHFFFAOYSA-N | $1.3\times10^1$ $3.5\times10^2$ $7.1\times10^3$ $1.7\times10^4$ | | Barcelo and Hennion (1997) Goodarzi et al. (2010) Modarresi et al. (2007) Maniere et al. (2011) | X Q Q ? | 567 568 67 12, 165 |
| etaconazole $C_{14}H_{15}Cl_2N_3O_2$ [60207-93-4] DWRKFAJEBUWTQM-UHFFFAOYSA-N | $7.9\times10^3$ | | MacBean (2012a) | ? | |
| triadimenol $C_{14}H_{18}ClN_3O_2$ [55219-65-3] BAZVSMNPJJMILC-UHFFFAOYSA-N | $7.6\times10^6$ $3.8\times10^6$ $4.5\times10^5$ | | HSDB (2015) Mackay et al. (2006d) Maniere et al. (2011) | V V ? | 241, 165 |
| hexaconazole $C_{14}H_{17}Cl_2N_3O$ [79983-71-4] STMIIPIFODONDC-UHFFFAOYSA-N | $2.8\times10^1$ $6.0\times10^1$ | | Barcelo and Hennion (1997) Goodarzi et al. (2010) | X Q | 567 568 |
| triadimefon $C_{14}H_{16}ClN_3O_2$ [43121-43-3] WURBVZBTWMNKQT-UHFFFAOYSA-N | $1.2\times10^5$ $1.2\times10^5$ $1.2\times10^5$ $1.5\times10^2$ | | Duchowicz et al. (2020) HSDB (2015) Mackay et al. (2006d) Duchowicz et al. (2020) | V V V Q | 186 |



Table A6.8: Chlorocarbons with nitrogen (C, H, O, N, Cl) (...continued)

| Substance Formula (Trivial Name) [CAS Registry Number] InChIKey | $H_s^{cp}$ (at $T^{\ominus}$) $\left[\dfrac{\mathrm{mol}}{\mathrm{m}^3\,\mathrm{Pa}}\right]$ | $\dfrac{\mathrm{d}\ln H_s^{cp}}{\mathrm{d}(1/T)}$ [K] | Reference | Type | Note |
|---|---|---|---|---|---|
| imazalil $C_{14}H_{14}Cl_2N_2O$ [35554-44-0] PZBPKYOVPCNPJY-UHFFFAOYSA-N | $3.8\times10^3$ $5.1\times10^5$ $9.3\times10^3$ | | HSDB (2015) Mackay et al. (2006d) Maniere et al. (2011) | V V ? | 241, 165 |
| pyrifenox $C_{14}H_{12}Cl_2N_2O$ [88283-41-4] CKPCAYZTYMHQEX-UHFFFAOYSA-N | $2.3\times10^2$ | | Ebert et al. (2023) | ? | 318 |
| chlorphthalim $C_{14}H_{12}ClNO_2$ [39985-63-2] MJQBFSWPMMHVSM-UHFFFAOYSA-N | $5.1\times10^2$ | | Ebert et al. (2023) | ? | 318 |
| cyprofuram $C_{14}H_{14}ClNO_3$ [69581-33-5] KRZUZYJEQBXUIN-UHFFFAOYSA-N | $3.1\times10^5$ | | Ebert et al. (2023) | ? | 316 |
| ofurace $C_{14}H_{16}ClNO_3$ [58810-48-3] OWDLFBLNMPCXSD-UHFFFAOYSA-N | $1.1\times10^4$ | | Ebert et al. (2023) | ? | 318 |
| triadimenol B $C_{14}H_{18}ClN_3O_2$ [82200-72-4] | $1.1\times10^5$ | | Ebert et al. (2023) | ? | 365 |
| triadimenol A $C_{14}H_{18}ClN_3O_2$ [89482-17-7] | $1.3\times10^5$ | | Ebert et al. (2023) | ? | 318 |
| propiconazole $C_{15}H_{17}Cl_2N_3O_2$ [60207-90-1] STJLVHWMYQXCPB-UHFFFAOYSA-N | $1.1\times10^4$ $5.7\times10^3$ $2.5\times10^3$ $2.4\times10^1$ $3.1\times10^1$ | | HSDB (2015) Mackay et al. (2006d) Siebers et al. (1994) Barcelo and Hennion (1997) Goodarzi et al. (2010) | V V V X Q | 567 568, 571 |
| clonazepam $C_{15}H_{10}ClN_3O_3$ [1622-61-3] DGBIGWXXNGSACT-UHFFFAOYSA-N | $1.4\times10^7$ | | HSDB (2015) | Q | 99 |
| oxazepam $C_{15}H_{12}ClN_2O_2$ [604-75-1] ADIMAYPTOBDMTL-UHFFFAOYSA-N | $1.8\times10^4$ | | HSDB (2015) | Q | 99 |



Table A6.8: Chlorocarbons with nitrogen (C, H, O, N, Cl) (...continued)

| Substance Formula (Trivial Name) [CAS Registry Number] InChIKey | $H_s^{cp}$ (at $T^{\ominus}$) $\left[\dfrac{\text{mol}}{\text{m}^3\,\text{Pa}}\right]$ | $\dfrac{\text{d}\ln H_s^{cp}}{\text{d}(1/T)}$ [K] | Reference | Type | Note |
|---|---|---|---|---|---|
| oxadiazon C$_{15}$H$_{18}$Cl$_2$N$_2$O$_3$ [19666-30-9] CHNUNORXWHYHNE-UHFFFAOYSA-N | $1.4\times10^2$ $1.4\times10^2$ | | HSDB (2015) Armbrust (2000) | V C | |
| cyproconazole C$_{15}$H$_{18}$ClN$_3$O [94361-06-5] UFNOUKDBUJZYDE-UHFFFAOYSA-N | $1.4\times10^4$ $1.3\times10^2$ $3.4\times10^2$ $2.0\times10^4$ | | HSDB (2015) Barcelo and Hennion (1997) Goodarzi et al. (2010) Maniere et al. (2011) | V X Q ? | 567 568 241, 165 |
| diclobutrazol C$_{15}$H$_{19}$Cl$_2$N$_3$O [75736-33-3] URDNHJIVMYZFRT-UHFFFAOYSA-N | $8.0\times10^3$ | | MacBean (2012a) | ? | |
| paclobutrazol C$_{15}$H$_{20}$ClN$_3$O [76738-62-0] RMOGWMIKYWRTKW-UHFFFAOYSA-N | $4.2\times10^4$ | | Maniere et al. (2011) | ? | 241, 165 |
| prochloraz C$_{15}$H$_{16}$Cl$_3$N$_3$O$_2$ [67747-09-5] TVLSRXXIMLFWEO-UHFFFAOYSA-N | $6.0\times10^2$ $5.8$ $6.4\times10^3$ $5.9\times10^{-1}$ $6.1\times10^2$ | | Duchowicz et al. (2020) Barcelo and Hennion (1997) Duchowicz et al. (2020) Goodarzi et al. (2010) Maniere et al. (2011) | V X Q Q ? | 186 567 568 241, 165 |
| *cis*-propiconazole C$_{15}$H$_{17}$Cl$_2$N$_3$O$_2$ [112721-87-6] | $1.2\times10^4$ | | Ebert et al. (2023) | ? | 318 |
| *trans*-propiconazole C$_{15}$H$_{17}$Cl$_2$N$_3$O$_2$ [120523-07-1] | $7.8\times10^3$ | | Ebert et al. (2023) | ? | 318 |
| fenoxanil C$_{15}$H$_{18}$Cl$_2$N$_2$O$_2$ [115852-48-7] IUOKJNROJISWRO-UHFFFAOYSA-N | $5.0\times10^6$ | | Ebert et al. (2023) | ? | 318 |
| carpropamid C$_{15}$H$_{18}$Cl$_3$NO [104030-54-8] RXDMAYSSBPYBFW-UHFFFAOYSA-N | $8.4\times10^2$ | | Ebert et al. (2023) | ? | 318 |
| propisochlor C$_{15}$H$_{22}$ClNO$_2$ [86763-47-5] KZNDFYDURHAESM-UHFFFAOYSA-N | $5.6\times10^1$ | | Ebert et al. (2023) | ? | 318 |



Table A6.8: Chlorocarbons with nitrogen (C, H, O, N, Cl) (...continued)

| Substance Formula (Trivial Name) [CAS Registry Number] InChIKey | $H_s^{cp}$ (at $T^\ominus$) $\left[\dfrac{\text{mol}}{\text{m}^3\,\text{Pa}}\right]$ | $\dfrac{\text{d}\ln H_s^{cp}}{\text{d}(1/T)}$ [K] | Reference | Type | Note |
|---|---|---|---|---|---|
| mefenpyr-diethyl $C_{16}H_{18}Cl_2N_2O_4$ [135590-91-9] OPGCOAPTHCZZIW-UHFFFAOYSA-N | $3.8\times10^3$ $8.5\times10^3$ | | Maniere et al. (2011) Maniere et al. (2011) | ? ? | 165 12, 165 |
| diazepam $C_{16}H_{13}ClN_2O$ [439-14-5] AAOVKJBEBIDNHE-UHFFFAOYSA-N | $2.7\times10^3$ | | HSDB (2015) | Q | 99 |
| bendamustine $C_{16}H_{21}Cl_2N_3O_2$ [16506-27-7] YTKUWDBFDASYHO-UHFFFAOYSA-N | $2.5\times10^7$ | | HSDB (2015) | Q | 99 |
| fentrazamide $C_{16}H_{20}ClN_5O_2$ [158237-07-1] LLQPHQFNMLZJMP-UHFFFAOYSA-N | $5.1\times10^4$ | | Ebert et al. (2023) | ? | 318 |
| piperalin $C_{16}H_{21}Cl_2NO_2$ [3478-94-2] BZGLBXYQOMFXAU-UHFFFAOYSA-N | $4.3\times10^2$ | | HSDB (2015) | Q | 99 |
| tebuconazole $C_{16}H_{22}ClN_3O$ [107534-96-3] PXMNMQRDXWABCY-UHFFFAOYSA-N | $7.0\times10^4$ $1.1\times10^3$ $2.2\times10^2$ $1.0\times10^5$ | | HSDB (2015) Barcelo and Hennion (1997) Goodarzi et al. (2010) Maniere et al. (2011) | V X Q ? | 567 568, 571 12, 165 |
| diethatyl-ethyl $C_{16}H_{22}ClNO_3$ [38727-55-8] WFKSADNZWSKCRZ-UHFFFAOYSA-N | $7.9\times10^2$ | | Ebert et al. (2023) | ? | 318 |
| fenarimol $C_{17}H_{12}Cl_2N_2O$ [60168-88-9] NHOWDZOIZKMVAI-UHFFFAOYSA-N | $1.4\times10^3$ $1.4\times10^3$ $7.8\times10^5$ | | Duchowicz et al. (2020) Mackay et al. (2006d) Duchowicz et al. (2020) | V V Q | 186 |
| cumyluron $C_{17}H_{19}ClN_2O$ [99485-76-4] VYNOULHXXDFBLU-UHFFFAOYSA-N | $2.5\times10^6$ | | Ebert et al. (2023) | ? | 318 |
| triticonazole $C_{17}H_{20}N_3OCl$ [131983-72-7] PPDBOQMNKNNODG-ZROIWOOFSA-N | $6.6\times10^4$ $3.3\times10^4$ | | HSDB (2015) Maniere et al. (2011) | Q ? | 99 241, 165 |



Table A6.8: Chlorocarbons with nitrogen (C, H, O, N, Cl) (...continued)

| Substance<br>Formula<br>(Trivial Name)<br>[CAS Registry Number]<br>InChIKey | $H_s^{cp}$<br>(at $T^\ominus$)<br>$\left[\dfrac{\text{mol}}{\text{m}^3\,\text{Pa}}\right]$ | $\dfrac{\text{d}\ln H_s^{cp}}{\text{d}(1/T)}$<br><br>[K] | Reference | Type | Note |
|---|---|---|---|---|---|
| furametpyr<br>$C_{17}H_{20}ClN_3O_2$<br>[123572-88-3]<br>NRTLIYOWLVMQBO-UHFFFAOYSA-N | $6.0\times10^6$ | | Ebert et al. (2023) | ? | 318 |
| metconazole<br>$C_{17}H_{22}ClN_3O$<br>[125116-23-6]<br>XWPZUHJBOLQNMN-UHFFFAOYSA-N | $3.8\times10^3$<br>$1.3\times10^3$<br>$4.5\times10^6$ | | Duchowicz et al. (2020)<br>Duchowicz et al. (2020)<br>Maniere et al. (2011) | V<br>Q<br>? | 186<br><br>12, 165 |
| tepraloxydim<br>$C_{17}H_{24}ClNO_4$<br>[149979-41-9]<br>IOYNQIMAUDJVEI-BMVIKAAMSA-N | $5.2\times10^4$ | | Ebert et al. (2023) | ? | 318 |
| cloquintocet-mexyl<br>$C_{18}H_{22}ClNO_3$<br>[99607-70-2]<br>COYBRKAVBMYYSF-UHFFFAOYSA-N | $1.2\times10^4$<br>$3.4\times10^2$<br>$3.3\times10^2$ | | Duchowicz et al. (2020)<br>Duchowicz et al. (2020)<br>Maniere et al. (2011) | V<br>Q<br>? | 186<br><br>165 |
| fenoxaprop-p-ethyl<br>$C_{18}H_{16}ClNO_5$<br>[71283-80-2]<br>PQKBPHSEKWERTG-LLVKDONJSA-N | $6.2\times10^2$<br>$2.3\times10^3$<br>$3.7\times10^3$ | | Duchowicz et al. (2020)<br>Duchowicz et al. (2020)<br>Maniere et al. (2011) | V<br>Q<br>? | 186<br><br>12, 165 |
| ipconazole<br>$C_{18}H_{24}ClN_3O$<br>[125225-28-7]<br>QTYCMDBMOLSEAM-UHFFFAOYSA-N | $3.9\times10^3$<br>$1.3\times10^3$<br>$3.3\times10^4$ | | Duchowicz et al. (2020)<br>Duchowicz et al. (2020)<br>Maniere et al. (2011) | V<br>Q<br>? | 186<br><br>241, 165 |
| pyriofenone<br>$C_{18}H_{20}ClNO_5$<br>[688046-61-9]<br>NMVCBWZLCXANER-UHFFFAOYSA-N | $5.3\times10^3$ | | Maniere et al. (2011) | ? | 241, 165 |
| cintofen<br><br>$C_{18}H_{15}ClN_2O_5$<br><br>(sintofen)<br><br>[130561-48-7]<br>QLMNCUHSDAGQGT-UHFFFAOYSA-N | $1.4\times10^7$<br><br>$5.4\times10^9$<br><br>$6.1\times10^9$ | | Maniere et al. (2011)<br><br>Maniere et al. (2011)<br><br>Maniere et al. (2011) | ?<br><br>?<br><br>? | 12, 577,<br>165<br>12, 573,<br>165<br>12, 493,<br>165 |
| boscalid<br>$C_{18}H_{12}Cl_2N_2O$<br>[188425-85-6]<br>WYEMLYFITZORAB-UHFFFAOYSA-N | $1.9\times10^4$<br>$1.9\times10^4$ | | MacBean (2012b)<br>Maniere et al. (2011) | X<br>? | 350<br>241, 165 |





Table A6.8: Chlorocarbons with nitrogen (C, H, O, N, Cl) (…continued)

| Substance<br>Formula<br>(Trivial Name)<br>[CAS Registry Number]<br>InChIKey | $H_s^{cp}$<br>(at $T^\ominus$)<br>$\left[\dfrac{\mathrm{mol}}{\mathrm{m^3\,Pa}}\right]$ | $\dfrac{\mathrm{d}\ln H_s^{cp}}{\mathrm{d}(1/T)}$<br><br>[K] | Reference | Type | Note |
|---|---|---|---|---|---|
| tebufenpyrad<br>$C_{18}H_{24}ClN_3O$<br>[119168-77-3]<br>ZZYSLNWGKKDOML-UHFFFAOYSA-N | $>8.2\times10^2$<br>$>9.1\times10^2$ | | HSDB (2015)<br>Maniere et al. (2011) | V<br>? | <br>241, 165 |
| 8,9,10,11-tetrachloro-12-<br>phthaloperinone<br>$C_{18}H_6Cl_4N_2O$<br>[20749-68-2]<br>UBZVRROHBDDCQY-UHFFFAOYSA-N | $4.8\times10^5$<br><br>$7.5\times10^9$<br>$5.7\times10^3$<br>$1.9\times10^5$ | | Zhang et al. (2010)<br><br>Zhang et al. (2010)<br>Zhang et al. (2010)<br>Zhang et al. (2010) | Q<br><br>Q<br>Q<br>Q | 287, 288<br><br>287, 289<br>287, 290<br>287, 291 |
| pigment red 254<br>$C_{18}H_{10}Cl_2N_2O_2$<br>[84632-65-5]<br>JNNHVXMCVRYTTN-UHFFFAOYSA-N | $3.4\times10^9$<br>$3.9\times10^6$<br>$1.9\times10^{13}$<br>$2.2\times10^{12}$ | | Zhang et al. (2010)<br>Zhang et al. (2010)<br>Zhang et al. (2010)<br>Zhang et al. (2010) | Q<br>Q<br>Q<br>Q | 287, 288<br>287, 289<br>287, 290<br>287, 291 |
| fenoxaprop-ethyl<br>$C_{18}H_{16}ClNO_5$<br>[66441-23-4]<br>PQKBPHSEKWERTG-UHFFFAOYSA-N | $6.2\times10^2$ | | Ebert et al. (2023) | ? | 316 |
| benzoylprop-ethyl<br>$C_{18}H_{17}Cl_2NO_3$<br>[22212-55-1]<br>SLCGUGMPSUYJAY-UHFFFAOYSA-N | $1.2\times10^4$ | | Ebert et al. (2023) | ? | 739 |
| difenoconazole<br>$C_{19}H_{17}Cl_2N_3O_3$<br>[119446-68-3]<br>BQYJATMQXGBDHF-UHFFFAOYSA-N | $1.1\times10^6$<br>$6.6\times10^3$<br>$3.1\times10^4$<br>$7.7\times10^1$<br>$1.1\times10^6$ | | Duchowicz et al. (2020)<br>Barcelo and Hennion (1997)<br>Duchowicz et al. (2020)<br>Goodarzi et al. (2010)<br>Maniere et al. (2011) | V<br>X<br>Q<br>Q<br>? | 186<br>567<br><br>568<br>165 |
| quizalofop ethyl<br>$C_{19}H_{17}ClN_2O_4$<br>[76578-14-8]<br>OSUHJPCHFDQAIT-UHFFFAOYSA-N | $9.3\times10^2$<br>$9.0\times10^2$<br>$4.0\times10^4$<br>$1.5\times10^4$ | | Duchowicz et al. (2020)<br>HSDB (2015)<br>Duchowicz et al. (2020)<br>Maniere et al. (2011) | V<br>V<br>Q<br>? | 186<br><br><br>12, 165 |
| quizalofop-p-ethyl<br>$C_{19}H_{17}ClN_2O_4$<br>[100646-51-3]<br>OSUHJPCHFDQAIT-GFCCVEGCSA-N | $4.8\times10^3$ | | Ebert et al. (2023) | ? | 318 |
| pyraclostrobine<br>$C_{19}H_{18}ClN_3O_4$<br>[175013-18-0]<br>HZRSNVGNWUDEFX-UHFFFAOYSA-N | $1.9\times10^5$<br>$1.9\times10^5$ | | MacBean (2012b)<br>Maniere et al. (2011) | X<br>? | 350<br>12, 165 |
| pyrimidifen<br>$C_{20}H_{28}ClN_3O_2$<br>[105779-78-0]<br>ITKAIUGKVKDENI-UHFFFAOYSA-N | $3.6\times10^4$ | | Ebert et al. (2023) | ? | 318 |



Table A6.8: Chlorocarbons with nitrogen (C, H, O, N, Cl) (...continued)

| Substance Formula (Trivial Name) [CAS Registry Number] InChIKey | $H_s^{cp}$ (at $T^\ominus$) $\left[\dfrac{\text{mol}}{\text{m}^3\,\text{Pa}}\right]$ | $\dfrac{\text{d}\ln H_s^{cp}}{\text{d}(1/T)}$ [K] | Reference | Type | Note |
|---|---|---|---|---|---|
| ponsol red violet 2rnx | $2.4\times10^{10}$ | | Zhang et al. (2010) | Q | 287, 288 |
| $C_{21}H_8Cl_3NO_3$ | $4.1\times10^7$ | | Zhang et al. (2010) | Q | 287, 289 |
| [6373-31-5] | $9.9\times10^8$ | | Zhang et al. (2010) | Q | 287, 290 |
| SQAQTWYUQXFOMH-UHFFFAOYSA-N | $4.6\times10^9$ | | Zhang et al. (2010) | Q | 287, 291 |
| dimethomorph | $4.9\times10^4$ | | Duchowicz et al. (2020) | V | 186 |
| $C_{21}H_{22}NO_4Cl$ | $9.3\times10^5$ | | Duchowicz et al. (2020) | Q | |
| [110488-70-5] | $9.9\times10^9$ | | HSDB (2015) | Q | 99 |
| QNBTYORWCCMPQP-JXAWBTAJSA-N | $1.9\times10^5$ | | Maniere et al. (2011) | ? | 241, 165 |
| | $4.0\times10^4$ | | Maniere et al. (2011) | ? | 241, 165 |
| propaquizafop | $3.2\times10^6$ | | Duchowicz et al. (2020) | V | 186 |
| $C_{22}H_{22}ClN_3O_5$ | $1.1\times10^7$ | | Duchowicz et al. (2020) | Q | |
| [111479-05-1] | $1.1\times10^7$ | | Maniere et al. (2011) | ? | 12, 165 |
| FROBCXTULYFHEJ-UHFFFAOYSA-N | | | | | |
| aripiprazole | $9.9\times10^{11}$ | | HSDB (2015) | Q | 99 |
| $C_{23}H_{27}Cl_2N_3O_2$ | | | | | |
| [129722-12-9] | | | | | |
| CEUORZQYGODEFX-UHFFFAOYSA-N | | | | | |
| ag-g-86814 | $1.8\times10^{14}$ | | Zhang et al. (2010) | Q | 287, 288 |
| $C_{26}H_6Cl_8N_2O_4$ | $9.7\times10^{12}$ | | Zhang et al. (2010) | Q | 287, 289 |
| [30125-47-4] | $1.1\times10^{11}$ | | Zhang et al. (2010) | Q | 287, 290 |
| ZEHOVWPIGREOPO-UHFFFAOYSA-N | $4.1\times10^{14}$ | | Zhang et al. (2010) | Q | 287, 291 |
| 8,18-dichloro-5,15-diethyl-5,15-dihydrodiindolo(3,2-b:3',2'-m)triphenodioxazine | $8.0\times10^6$ | | Zhang et al. (2010) | Q | 287, 288 |
| $C_{34}H_{22}Cl_2N_4O_2$ | $1.8\times10^{12}$ | | Zhang et al. (2010) | Q | 287, 289 |
| [6358-30-1] | $6.2\times10^6$ | | Zhang et al. (2010) | Q | 287, 290 |
| CGLVZFOCZLHKOH-UHFFFAOYSA-N | $1.0\times10^{10}$ | | Zhang et al. (2010) | Q | 287, 291 |



### A6.9 Chlorofluorocarbons (C, H, O, N, F, Cl)

Table A6.9: Chlorofluorocarbons (C, H, O, N, F, Cl)

| Substance Formula (Trivial Name) [CAS Registry Number] InChIKey | $H_s^{cp}$ (at $T^{\ominus}$) $\left[\dfrac{\mathrm{mol}}{\mathrm{m^3\,Pa}}\right]$ | $\dfrac{\mathrm{d}\ln H_s^{cp}}{\mathrm{d}(1/T)}$ [K] | Reference | Type | Note |
|---|---|---|---|---|---|
| chlorofluoromethane | $1.5\times10^{-3}$ | 2600 | Wilhelm et al. (1977) | L | |
| CH$_2$FCl | $1.5\times10^{-3}$ | 2300 | Boggs and Buck (1958) | M | |
| (R31) | $1.5\times10^{-3}$ | | Duchowicz et al. (2020) | V | 186 |
| [593-70-4] | $1.5\times10^{-3}$ | | Hine and Mookerjee (1975) | V | |
| XWCDCDSDNJVCLO-UHFFFAOYSA-N | $1.6\times10^{-3}$ | | Yaws (2003) | X | 237 |
| | $1.5\times10^{-3}$ | | Duchowicz et al. (2020) | Q | |
| | $3.9\times10^{-4}$ | | Gharagheizi et al. (2012) | Q | |
| | $1.2\times10^{-3}$ | | Raventos-Duran et al. (2010) | Q | 242, 243 |
| | $3.9\times10^{-3}$ | | Raventos-Duran et al. (2010) | Q | 244 |
| | $2.0\times10^{-4}$ | | Raventos-Duran et al. (2010) | Q | 245 |
| | $1.6\times10^{-3}$ | | Gharagheizi et al. (2010) | Q | 246 |
| | $3.4\times10^{-3}$ | | Hilal et al. (2008) | Q | |
| | $5.9\times10^{-4}$ | | Modarresi et al. (2007) | Q | 67 |
| | | 2600 | Kühne et al. (2005) | Q | |
| | $6.6\times10^{-4}$ | | Yao et al. (2002) | Q | 229 |
| | $1.8\times10^{-3}$ | | English and Carroll (2001) | Q | 230, 231 |
| | $6.1\times10^{-4}$ | | Nirmalakhandan and Speece (1988) | Q | |
| | $1.8\times10^{-3}$ | | Irmann (1965) | Q | |
| | | 2500 | Kühne et al. (2005) | ? | |
| | $1.6\times10^{-3}$ | | Yaws (1999) | ? | 21 |
| | $1.5\times10^{-3}$ | | Yaws and Yang (1992) | ? | 21 |
| chlorodifluoromethane | $3.5\times10^{-4}$ | 2600 | Burkholder et al. (2019) | L | |
| CHF$_2$Cl | $3.2\times10^{-4}$ | 2700 | Burkholder et al. (2019) | L | 70 |
| (R22) | $3.5\times10^{-4}$ | 2600 | Burkholder et al. (2015) | L | |
| [75-45-6] | $3.2\times10^{-4}$ | 2700 | Burkholder et al. (2015) | L | 70 |
| VOPWNXZWBYDODV-UHFFFAOYSA-N | $3.4\times10^{-4}$ | 3400 | Sander et al. (2011) | L | 1 |
| | $3.4\times10^{-4}$ | 3400 | Wilhelm et al. (1977) | L | |
| | $2.9\times10^{-4}$ | 3100 | Ooki and Yokouchi (2011) | M | 70 |
| | $3.4\times10^{-4}$ | 3000 | Zheng et al. (1997) | M | 740 |
| | $3.5\times10^{-4}$ | 3100 | Maaßen (1995) | M | 741 |
| | $3.5\times10^{-4}$ | 3000 | Reichl (1995) | M | 742 |
| | $1.7\times10^{-4}$ | 3500 | Chang and Criddle (1995) | M | 743 |
| | $3.5\times10^{-4}$ | 2600 | Boggs and Buck (1958) | M | |
| | $3.3\times10^{-4}$ | | Mackay et al. (2006b) | V | |
| | $3.3\times10^{-4}$ | | Mackay et al. (1993) | V | |
| | $3.3\times10^{-4}$ | 3600 | McLinden (1989) | V | 744, 745 |
| | $3.4\times10^{-4}$ | | Hine and Mookerjee (1975) | V | |
| | $3.2\times10^{-4}$ | | Irmann (1965) | V | |
| | $3.3\times10^{-4}$ | | Yaws (2003) | X | 237 |
| | | | Kanakidou et al. (1995) | C | 746 |
| | $3.1\times10^{-4}$ | | Hayer et al. (2022) | Q | 20 |
| | $3.4\times10^{-4}$ | 3100 | Li et al. (2019) | Q | 1 |
| | $1.4\times10^{-4}$ | | Gharagheizi et al. (2012) | Q | |
| | $3.9\times10^{-4}$ | | Raventos-Duran et al. (2010) | Q | 242, 243 |



Table A6.9: Chlorofluorocarbons (C, H, O, N, F, Cl) (...continued)

| Substance Formula (Trivial Name) [CAS Registry Number] InChIKey | $H_s^{cp}$ (at $T^{\ominus}$) $\left[\dfrac{\mathrm{mol}}{\mathrm{m^3\,Pa}}\right]$ | $\dfrac{\mathrm{d}\ln H_s^{cp}}{\mathrm{d}(1/T)}$ [K] | Reference | Type | Note |
|---|---|---|---|---|---|
| | $6.2\times10^{-4}$ | | Raventos-Duran et al. (2010) | Q | 244 |
| | $9.9\times10^{-5}$ | | Raventos-Duran et al. (2010) | Q | 245 |
| | $3.7\times10^{-4}$ | | Gharagheizi et al. (2010) | Q | 246 |
| | $6.0\times10^{-4}$ | | Hilal et al. (2008) | Q | |
| | $1.9\times10^{-4}$ | | Modarresi et al. (2007) | Q | 67 |
| | | 2600 | Kühne et al. (2005) | Q | |
| | $3.4\times10^{-4}$ | | Yaffe et al. (2003) | Q | 248, 249 |
| | $8.4\times10^{-4}$ | | English and Carroll (2001) | Q | 230, 231 |
| | $1.0\times10^{-4}$ | | Katritzky et al. (1998) | Q | |
| | $4.0\times10^{-4}$ | | Nirmalakhandan and Speece (1988) | Q | |
| | $3.5\times10^{-4}$ | | Irmann (1965) | Q | |
| | | 3000 | Kühne et al. (2005) | ? | |
| | $3.3\times10^{-4}$ | | Yaws (1999) | ? | 21 |
| | $2.4\times10^{-4}$ | | Abraham and Weathersby (1994) | ? | 21 |
| | $3.3\times10^{-4}$ | | Yaws and Yang (1992) | ? | 21 |
| dichlorofluoromethane CHFCl$_2$ (R21) [75-43-4] UMNKXPULIDJLSU-UHFFFAOYSA-N | $9.1\times10^{-4}$ | | HSDB (2015) | V | |
| | $1.8\times10^{-3}$ | | Mackay et al. (1993) | V | |
| | $6.8\times10^{-4}$ | | Gharagheizi et al. (2012) | Q | |
| | $1.6\times10^{-3}$ | | Hilal et al. (2008) | Q | |
| | $5.9\times10^{-4}$ | | Modarresi et al. (2007) | Q | 67 |
| | $9.2\times10^{-4}$ | | Yaffe et al. (2003) | Q | 248, 249 |
| | $1.5\times10^{-3}$ | | Yao et al. (2002) | Q | 229 |
| | $7.2\times10^{-4}$ | | Katritzky et al. (1998) | Q | |
| | $1.9\times10^{-3}$ | | Yaws (1999) | ? | 21 |
| | $3.8\times10^{-5}$ | | Mackay et al. (1993) | ? | |
| | $1.9\times10^{-3}$ | | Yaws and Yang (1992) | ? | 21 |
| chlorotrifluoromethane CF$_3$Cl (R13) [75-72-9] AFYPFACVUDMOHA-UHFFFAOYSA-N | $9.9\times10^{-6}$ | 1700 | Burkholder et al. (2019) | L | 747 |
| | $8.0\times10^{-6}$ | 1500 | Burkholder et al. (2019) | L | 70 |
| | $9.9\times10^{-6}$ | 1700 | Burkholder et al. (2015) | L | 748 |
| | $8.0\times10^{-6}$ | 1500 | Burkholder et al. (2015) | L | 70 |
| | $9.9\times10^{-6}$ | 1700 | Sander et al. (2011) | L | 749 |
| | $9.3\times10^{-6}$ | 1600 | Wilhelm et al. (1977) | L | |
| | $8.9\times10^{-6}$ | 1900 | Reichl (1995) | M | 750 |
| | $9.0\times10^{-6}$ | 2100 | Scharlin and Battino (1995) | M | 751 |
| | $9.0\times10^{-6}$ | 2100 | Scharlin and Battino (1994) | M | 752 |
| | $7.8\times10^{-6}$ | | Park et al. (1982) | M | |
| | $1.5\times10^{-4}$ | | Mackay et al. (1993) | V | |
| | $5.7\times10^{-6}$ | | Hine and Mookerjee (1975) | V | |
| | $8.8\times10^{-6}$ | | Yaws (2003) | X | 237 |
| | $7.2\times10^{-6}$ | | Hilal et al. (2008) | C | |
| | $5.7\times10^{-6}$ | | Irmann (1965) | C | |
| | $8.3\times10^{-6}$ | | Hayer et al. (2022) | Q | 20 |
| | $6.4\times10^{-6}$ | | Keshavarz et al. (2022) | Q | |
| | $4.0\times10^{-5}$ | | Duchowicz et al. (2020) | Q | |
| | $2.3\times10^{-5}$ | | Gharagheizi et al. (2012) | Q | |
| | $6.9\times10^{-6}$ | | Gharagheizi et al. (2010) | Q | 246 |



Table A6.9: Chlorofluorocarbons (C, H, O, N, F, Cl) (. . . continued)

| Substance<br>Formula<br>(Trivial Name)<br>[CAS Registry Number]<br>InChIKey | $H_s^{cp}$<br>(at $T^{\ominus}$)<br><br>$\left[\dfrac{\text{mol}}{\text{m}^3\,\text{Pa}}\right]$ | $\dfrac{\text{d}\ln H_s^{cp}}{\text{d}(1/T)}$<br><br><br>[K] | Reference | Type | Note |
|---|---|---|---|---|---|
| | $2.6\times10^{-5}$ | | Hilal et al. (2008) | Q | |
| | $2.1\times10^{-5}$ | | Modarresi et al. (2007) | Q | 67 |
| | | 2600 | Kühne et al. (2005) | Q | |
| | $7.7\times10^{-6}$ | | Yaffe et al. (2003) | Q | 248, 249 |
| | $6.7\times10^{-7}$ | | Katritzky et al. (1998) | Q | |
| | $1.4\times10^{-5}$ | | Nirmalakhandan and Speece (1988) | Q | |
| | $5.1\times10^{-6}$ | | Irmann (1965) | Q | |
| | $7.2\times10^{-6}$ | | Duchowicz et al. (2020) | ? | 185, 21 |
| | | 2000 | Kühne et al. (2005) | ? | |
| | $8.8\times10^{-6}$ | | Yaws (1999) | ? | 21 |
| | $8.7\times10^{-6}$ | | Yaws and Yang (1992) | ? | 21 |
| dichlorodifluoromethane | $3.0\times10^{-5}$ | 3500 | Burkholder et al. (2019) | L | |
| $CF_2Cl_2$ | $2.6\times10^{-5}$ | 2100 | Burkholder et al. (2019) | L | 70 |
| (R12) | $3.0\times10^{-5}$ | 3500 | Burkholder et al. (2015) | L | |
| [75-71-8] | $2.6\times10^{-5}$ | 2100 | Burkholder et al. (2015) | L | 70 |
| PXBRQCKWGAHEHS-UHFFFAOYSA-N | $3.1\times10^{-5}$ | 3200 | Brockbank (2013) | L | 1 |
| | $3.0\times10^{-5}$ | 3400 | Warneck and Williams (2012) | L | |
| | $3.0\times10^{-5}$ | 3500 | Sander et al. (2011) | L | |
| | $3.0\times10^{-5}$ | 3500 | Sander et al. (2006) | L | |
| | $3.1\times10^{-5}$ | 3500 | Staudinger and Roberts (2001) | L | |
| | $2.1\times10^{-5}$ | 1800 | Wilhelm et al. (1977) | L | |
| | $1.3\times10^{-4}$ | 5500 | Hiatt (2013) | M | |
| | $3.0\times10^{-5}$ | 3000 | Reichl (1995) | M | 753 |
| | $2.9\times10^{-5}$ | 2900 | Scharlin and Battino (1995) | M | 754 |
| | $2.9\times10^{-5}$ | 2900 | Scharlin and Battino (1994) | M | 755 |
| | $3.1\times10^{-5}$ | 3500 | Munz and Roberts (1987) | M | |
| | $2.9\times10^{-5}$ | 3200 | Warner and Weiss (1985) | M | |
| | $2.3\times10^{-5}$ | 3400 | Wisegarver and Cline (1985) | M | 70 |
| | $2.9\times10^{-5}$ | | Park et al. (1982) | M | |
| | $2.5\times10^{-5}$ | | Pearson and McConnell (1975) | M | 649, 12 |
| | $2.4\times10^{-5}$ | | Mackay et al. (2006b) | V | |
| | $2.4\times10^{-5}$ | | Mackay et al. (1993) | V | |
| | $2.3\times10^{-5}$ | | Mackay and Shiu (1981) | V | |
| | $2.3\times10^{-5}$ | | Hine and Mookerjee (1975) | V | |
| | $2.5\times10^{-5}$ | | Yaws (2003) | X | 237 |
| | $3.5\times10^{-6}$ | -210 | Goldstein (1982) | X | 298 |
| | $3.6\times10^{-5}$ | | Hilal et al. (2008) | C | |
| | $6.4\times10^{-6}$ | | Ryan et al. (1988) | C | |
| | $2.3\times10^{-5}$ | | Irmann (1965) | C | |
| | $2.4\times10^{-5}$ | | Hayer et al. (2022) | Q | 20 |
| | $1.9\times10^{-5}$ | | Keshavarz et al. (2022) | Q | |
| | $1.1\times10^{-4}$ | | Duchowicz et al. (2020) | Q | 184 |
| | $2.8\times10^{-5}$ | 3300 | Li et al. (2019) | Q | 1 |
| | $1.2\times10^{-4}$ | | Gharagheizi et al. (2012) | Q | |
| | $2.4\times10^{-5}$ | | Gharagheizi et al. (2010) | Q | 246 |
| | $5.4\times10^{-5}$ | | Hilal et al. (2008) | Q | |
| | $7.7\times10^{-5}$ | | Modarresi et al. (2007) | Q | 67 |





Table A6.9: Chlorofluorocarbons (C, H, O, N, F, Cl) (... continued)

| Substance<br>Formula<br>(Trivial Name)<br>[CAS Registry Number]<br>InChIKey | $H_s^{cp}$<br>(at $T^{\ominus}$)<br>$\left[\dfrac{\mathrm{mol}}{\mathrm{m}^3\,\mathrm{Pa}}\right]$ | $\dfrac{\mathrm{d}\ln H_s^{cp}}{\mathrm{d}(1/T)}$<br><br>[K] | Reference | Type | Note |
|---|---|---|---|---|---|
| | | 3000 | Kühne et al. (2005) | Q | |
| | $2.5\times10^{-5}$ | | Yaffe et al. (2003) | Q | 248, 249 |
| | $1.1\times10^{-5}$ | | Katritzky et al. (1998) | Q | |
| | $4.7\times10^{-5}$ | | Nirmalakhandan and Speece (1988) | Q | |
| | $2.0\times10^{-5}$ | | Irmann (1965) | Q | |
| | $2.9\times10^{-5}$ | | Duchowicz et al. (2020) | ? | 185, 21 |
| | | 3400 | Kühne et al. (2005) | ? | |
| | $2.5\times10^{-5}$ | | Yaws (1999) | ? | 21 |
| | $2.3\times10^{-5}$ | | Abraham and Weathersby (1994) | ? | 21 |
| | $2.5\times10^{-5}$ | | Yaws and Yang (1992) | ? | 21 |
| trichlorofluoromethane | $1.1\times10^{-4}$ | 3300 | Burkholder et al. (2019) | L | |
| CFCl$_3$ | $7.3\times10^{-5}$ | 3900 | Burkholder et al. (2019) | L | 70 |
| (R11) | $1.1\times10^{-4}$ | 3300 | Burkholder et al. (2015) | L | |
| [75-69-4] | $7.3\times10^{-5}$ | 3900 | Burkholder et al. (2015) | L | 70 |
| CYRMSUTZVYGINF-UHFFFAOYSA-N | $1.0\times10^{-4}$ | 3400 | Brockbank (2013) | L | 1 |
| | $1.1\times10^{-4}$ | 3400 | Warneck and Williams (2012) | L | |
| | $1.1\times10^{-4}$ | 3300 | Sander et al. (2011) | L | |
| | $1.1\times10^{-4}$ | 3300 | Sander et al. (2006) | L | |
| | $1.1\times10^{-4}$ | 3300 | Staudinger and Roberts (2001) | L | |
| | $1.0\times10^{-4}$ | 3100 | Staudinger and Roberts (1996) | L | |
| | $2.8\times10^{-4}$ | 5100 | Hiatt (2013) | M | |
| | $6.0\times10^{-5}$ | 4900 | Ooki and Yokouchi (2011) | M | 70 |
| | $1.0\times10^{-4}$ | 3700 | Maaßen (1995) | M | 756 |
| | $1.5\times10^{-4}$ | 3700 | Reichl (1995) | M | 757 |
| | $9.8\times10^{-5}$ | 3500 | Ashworth et al. (1988) | M | 278 |
| | $1.0\times10^{-4}$ | 3600 | Warner and Weiss (1985) | M | |
| | $7.8\times10^{-5}$ | 3900 | Wisegarver and Cline (1985) | M | 70 |
| | $1.1\times10^{-4}$ | 2700 | Hunter-Smith et al. (1983) | M | 658 |
| | $1.1\times10^{-4}$ | | Park et al. (1982) | M | |
| | $1.7\times10^{-4}$ | | Warner et al. (1980) | M | |
| | $1.1\times10^{-4}$ | 2100 | Balls (1980) | M | |
| | $1.2\times10^{-5}$ | | Pearson and McConnell (1975) | M | 649, 12 |
| | $7.8\times10^{-5}$ | | Mackay et al. (2006b) | V | |
| | $9.9\times10^{-5}$ | 6100 | Fogg and Sangster (2003) | V | |
| | $7.8\times10^{-5}$ | | Mackay et al. (1993) | V | |
| | $9.0\times10^{-5}$ | | Yoshida et al. (1983) | V | |
| | $9.0\times10^{-5}$ | | Mackay and Shiu (1981) | V | |
| | $9.5\times10^{-5}$ | | Warner et al. (1980) | V | |
| | $9.8\times10^{-5}$ | | Irmann (1965) | V | |
| | $8.0\times10^{-5}$ | | Yaws (2003) | X | 237, 38 |
| | $1.7\times10^{-4}$ | 730 | Goldstein (1982) | X | 298 |
| | $1.0\times10^{-4}$ | | Hilal et al. (2008) | C | |
| | $1.7\times10^{-4}$ | | Ryan et al. (1988) | C | |
| | $1.7\times10^{-4}$ | | Shen (1982) | C | |
| | $8.1\times10^{-5}$ | | Liss and Slater (1974) | C | |
| | $9.4\times10^{-5}$ | | Hayer et al. (2022) | Q | 20 |
| | $5.9\times10^{-5}$ | | Keshavarz et al. (2022) | Q | |



Table A6.9: Chlorofluorocarbons (C, H, O, N, F, Cl) (... continued)

| Substance<br>Formula<br>(Trivial Name)<br>[CAS Registry Number]<br>InChIKey | $H_s^{cp}$ <br>(at $T^{\ominus}$) <br>$\left[\dfrac{\mathrm{mol}}{\mathrm{m^3\,Pa}}\right]$ | $\dfrac{\mathrm{d}\ln H_s^{cp}}{\mathrm{d}(1/T)}$ <br>[K] | Reference | Type | Note |
|---|---|---|---|---|---|
| | $2.7\times10^{-4}$ | | Duchowicz et al. (2020) | Q | 299 |
| | $6.5\times10^{-4}$ | | Gharagheizi et al. (2012) | Q | |
| | $8.4\times10^{-5}$ | | Gharagheizi et al. (2010) | Q | 246 |
| | $1.7\times10^{-4}$ | | Hilal et al. (2008) | Q | |
| | $2.5\times10^{-4}$ | | Modarresi et al. (2007) | Q | 67 |
| | | 3300 | Kühne et al. (2005) | Q | |
| | $1.1\times10^{-4}$ | | Yaffe et al. (2003) | Q | 248, 249 |
| | $8.6\times10^{-5}$ | | Katritzky et al. (1998) | Q | |
| | $8.6\times10^{-5}$ | | Irmann (1965) | Q | |
| | $1.0\times10^{-4}$ | | Duchowicz et al. (2020) | ? | 185, 21 |
| | $9.8\times10^{-5}$ | | Mackay et al. (2006b) | ? | |
| | | 3800 | Kühne et al. (2005) | ? | |
| | $8.1\times10^{-5}$ | | Yaws (1999) | ? | 21, 38 |
| | $8.2\times10^{-5}$ | | Abraham and Weathersby (1994) | ? | 21 |
| | $9.8\times10^{-5}$ | | Mackay et al. (1993) | ? | |
| | $8.1\times10^{-5}$ | | Yaws and Yang (1992) | ? | 21 |
| 1,1,1,2-tetrachlorodifluoroethane<br>$C_2Cl_4F_2$<br>[76-11-9]<br>SLGOCMATMKJJCE-UHFFFAOYSA-N | $6.1\times10^{-5}$ | | Duchowicz et al. (2020) | V | 186 |
| | $6.2\times10^{-5}$ | | HSDB (2015) | V | |
| | $7.3\times10^{-4}$ | | Duchowicz et al. (2020) | Q | |
| | $5.1\times10^{-4}$ | | Hilal et al. (2008) | Q | |
| | $2.8\times10^{-4}$ | | Modarresi et al. (2007) | Q | 67 |
| | $2.4\times10^{-3}$ | | Yaffe et al. (2003) | Q | 248, 249 |
| | $3.2\times10^{-5}$ | | Katritzky et al. (1998) | Q | |
| 1,1,2,2-tetrachlorodifluoroethane<br>$C_2F_2Cl_4$<br>(R112)<br>[76-12-0]<br>UGCSPKPEHQEOSR-UHFFFAOYSA-N | $9.0\times10^{-5}$ | | HSDB (2015) | V | |
| | $1.0\times10^{-4}$ | | Hine and Mookerjee (1975) | V | |
| | $7.9\times10^{-5}$ | | Yaws (2003) | X | 237 |
| | $2.4\times10^{-3}$ | | Gharagheizi et al. (2012) | Q | |
| | $7.1\times10^{-5}$ | | Gharagheizi et al. (2010) | Q | 246 |
| | $5.1\times10^{-4}$ | | Hilal et al. (2008) | Q | |
| | $3.2\times10^{-4}$ | | Modarresi et al. (2007) | Q | 67 |
| | $1.1\times10^{-4}$ | | Yao et al. (2002) | Q | 229 |
| | $7.9\times10^{-5}$ | | Yaws (1999) | ? | 21 |
| 1,1,1-trichloro-2,2,2-<br>trifluoroethane<br>$C_2F_3Cl_3$<br>(R113a)<br>[354-58-5]<br>BOSAWIQFTJIYIS-UHFFFAOYSA-N | $3.4\times10^{-5}$ | 3200 | Burkholder et al. (2019) | L | |
| | $3.4\times10^{-5}$ | 3200 | Burkholder et al. (2015) | L | |
| | $3.7\times10^{-5}$ | | HSDB (2015) | Q | 99 |
| | $3.7\times10^{-5}$ | | Zhang et al. (2010) | Q | 287, 288 |
| | $2.1\times10^{-4}$ | | Zhang et al. (2010) | Q | 287, 289 |
| | $5.8\times10^{-5}$ | | Zhang et al. (2010) | Q | 287, 290 |
| | $3.0\times10^{-5}$ | | Zhang et al. (2010) | Q | 287, 291 |





Table A6.9: Chlorofluorocarbons (C, H, O, N, F, Cl) (. . . continued)

| Substance Formula (Trivial Name) [CAS Registry Number] InChIKey | $H_s^{cp}$ (at $T^{\ominus}$) $\left[\dfrac{\mathrm{mol}}{\mathrm{m^3\,Pa}}\right]$ | $\dfrac{\mathrm{d}\ln H_s^{cp}}{\mathrm{d}(1/T)}$ [K] | Reference | Type | Note |
|---|---|---|---|---|---|
| 1,1,2-trichloro-1,2,2-trifluoroethane | $2.9\times10^{-5}$ | 4100 | Brockbank (2013) | L | 1 |
| $C_2F_3Cl_3$ | $2.0\times10^{-4}$ | 5700 | Hiatt (2013) | M | |
| (R113) | $2.9\times10^{-5}$ | 4300 | Dewulf et al. (1999) | M | |
| [76-13-1] | $2.9\times10^{-5}$ | 4000 | Bu and Warner (1995) | M | 758 |
| AJDIZQLSFPQPEY-UHFFFAOYSA-N | $2.8\times10^{-5}$ | 6500 | Reichl (1995) | M | 759 |
| | $3.4\times10^{-5}$ | 3200 | Ashworth et al. (1988) | M | 278 |
| | $1.9\times10^{-5}$ | | HSDB (2015) | V | |
| | | | Mackay et al. (2006b) | V | 683 |
| | $8.8\times10^{-6}$ | | Mackay et al. (1993) | V | |
| | $2.0\times10^{-5}$ | | Hine and Mookerjee (1975) | V | |
| | $2.0\times10^{-5}$ | | Yaws (2003) | X | 237 |
| | $7.0\times10^{-4}$ | | Gharagheizi et al. (2012) | Q | |
| | $2.5\times10^{-5}$ | | Gharagheizi et al. (2010) | Q | 246 |
| | $1.8\times10^{-4}$ | | Hilal et al. (2008) | Q | |
| | $1.1\times10^{-4}$ | | Modarresi et al. (2007) | Q | 67 |
| | | 3700 | Kühne et al. (2005) | Q | |
| | $3.1\times10^{-5}$ | | Mackay et al. (2006b) | ? | |
| | | 3800 | Kühne et al. (2005) | ? | |
| | $2.1\times10^{-5}$ | | Yaws (1999) | ? | 21 |
| | $3.1\times10^{-5}$ | | Mackay et al. (1993) | ? | |
| | $2.0\times10^{-5}$ | | Yaws and Yang (1992) | ? | 21 |
| | $2.0\times10^{-5}$ | | Abraham et al. (1990) | ? | |
| 1,1-dichlorotetrafluoroethane | $8.2\times10^{-6}$ | | HSDB (2015) | V | |
| $C_2F_4Cl_2$ | $5.8\times10^{-6}$ | | Hine and Mookerjee (1975) | V | |
| (R114a) | $7.5\times10^{-6}$ | | Hilal et al. (2008) | C | |
| [374-07-2] | $5.8\times10^{-6}$ | | Irmann (1965) | C | 294 |
| BAMUEXIPKSRTBS-UHFFFAOYSA-N | $8.8\times10^{-5}$ | | Hilal et al. (2008) | Q | |
| | $6.6\times10^{-6}$ | | Irmann (1965) | Q | |
| 1,2-dichlorotetrafluoroethane | $9.0\times10^{-6}$ | 2800 | Reichl (1995) | M | 760 |
| $C_2F_4Cl_2$ | $7.9\times10^{-6}$ | | Mackay et al. (1993) | V | |
| (R114) | $8.0\times10^{-6}$ | | Hine and Mookerjee (1975) | V | |
| [76-14-2] | $8.1\times10^{-6}$ | | Yaws (2003) | X | 237 |
| DDMOUSALMHHKOS-UHFFFAOYSA-N | $8.1\times10^{-6}$ | | Irmann (1965) | C | 12 |
| | $7.1\times10^{-4}$ | | Hayer et al. (2022) | Q | 20 |
| | $1.8\times10^{-4}$ | | Gharagheizi et al. (2012) | Q | |
| | $8.1\times10^{-6}$ | | Gharagheizi et al. (2010) | Q | 246 |
| | $8.4\times10^{-5}$ | | Hilal et al. (2008) | Q | |
| | $2.8\times10^{-5}$ | | Modarresi et al. (2007) | Q | 67 |
| | | 3300 | Kühne et al. (2005) | Q | |
| | $1.1\times10^{-4}$ | | Yao et al. (2002) | Q | 229 |
| | $6.6\times10^{-6}$ | | Irmann (1965) | Q | |
| | | 2700 | Kühne et al. (2005) | ? | |
| | $8.2\times10^{-6}$ | | Yaws (1999) | ? | 21 |
| | $7.9\times10^{-6}$ | | Abraham and Weathersby (1994) | ? | 21 |
| | $8.1\times10^{-6}$ | | Yaws and Yang (1992) | ? | 21 |



Table A6.9: Chlorofluorocarbons (C, H, O, N, F, Cl) (...continued)

| Substance Formula (Trivial Name) [CAS Registry Number] InChIKey | $H_s^{cp}$ (at $T^{\ominus}$) $\left[\dfrac{\mathrm{mol}}{\mathrm{m^3\,Pa}}\right]$ | $\dfrac{\mathrm{d}\ln H_s^{cp}}{\mathrm{d}(1/T)}$ [K] | Reference | Type | Note |
|---|---|---|---|---|---|
| chloropentafluoroethane | $3.4\times10^{-6}$ | 2800 | Wilhelm et al. (1977) | L | |
| $C_2F_5Cl$ | $3.1\times10^{-6}$ | 2100 | Reichl (1995) | M | 761 |
| (R115) | $1.8\times10^{-6}$ | | Duchowicz et al. (2020) | V | 186 |
| [76-15-3] | $1.8\times10^{-6}$ | | HSDB (2015) | V | |
| RFCAUADVODFSLZ-UHFFFAOYSA-N | $3.8\times10^{-6}$ | | Mackay et al. (1993) | V | |
| | $3.7\times10^{-6}$ | | Meylan and Howard (1991) | V | |
| | $3.2\times10^{-6}$ | | Hine and Mookerjee (1975) | V | |
| | $3.8\times10^{-6}$ | | Yaws (2003) | X | 237 |
| | $3.2\times10^{-6}$ | | Irmann (1965) | C | |
| | $3.1\times10^{-6}$ | | Hayer et al. (2022) | Q | 20 |
| | $6.5\times10^{-5}$ | | Duchowicz et al. (2020) | Q | |
| | $4.9\times10^{-5}$ | | Gharagheizi et al. (2012) | Q | |
| | $2.5\times10^{-6}$ | | Gharagheizi et al. (2010) | Q | 246 |
| | $3.4\times10^{-5}$ | | Hilal et al. (2008) | Q | |
| | $7.6\times10^{-6}$ | | Modarresi et al. (2007) | Q | 67 |
| | | 2900 | Kühne et al. (2005) | Q | |
| | $2.4\times10^{-5}$ | | Yao et al. (2002) | Q | 229 |
| | $1.2\times10^{-6}$ | | Meylan and Howard (1991) | Q | |
| | $2.1\times10^{-6}$ | | Irmann (1965) | Q | |
| | | 2000 | Kühne et al. (2005) | ? | |
| | $3.8\times10^{-6}$ | | Yaws (1999) | ? | 21 |
| | $3.8\times10^{-6}$ | | Yaws and Yang (1992) | ? | 21 |
| 1,1,2,2-tetrachloro-1-fluoroethane $C_2HCl_4F$ [354-14-3] LUBCGHUOCJOIJA-UHFFFAOYSA-N | $3.3\times10^{-3}$ | | HSDB (2015) | Q | 99 |
| 1,1-dichloro-1,2,2-trifluoroethane $C_2HCl_2F_3$ [812-04-4] AFTSHZRGGNMLHC-UHFFFAOYSA-N | $1.0\times10^{-4}$ | | HSDB (2015) | Q | 99 |
| 1,2-dichloro-1,1,2-trifluoroethane $C_2HCl_2F_3$ [354-23-4] YMRMDGSNYHCUCL-UHFFFAOYSA-N | $1.0\times10^{-4}$ | | HSDB (2015) | Q | 99 |
| 2,2-dichloro-1,1,1-trifluoroethane | $2.3\times10^{-4}$ | 2400 | Kutsuna (2013) | M | |
| $C_2HF_3Cl_2$ | $3.2\times10^{-4}$ | 3100 | Chang and Criddle (1995) | M | 762 |
| (R123) | $2.8\times10^{-4}$ | 2600 | McLinden (1989) | V | |
| [306-83-2] | $5.0\times10^{-4}$ | | Hilal et al. (2008) | Q | |
| OHMHBGPWCHTMQE-UHFFFAOYSA-N | $1.8\times10^{-4}$ | | Modarresi et al. (2007) | Q | 67 |
| 1-chloro-1,1,2,2-tetrafluoroethane $C_2HClF_4$ [354-25-6] JQZFYIGAYWLRCC-UHFFFAOYSA-N | $1.8\times10^{-5}$ | | HSDB (2015) | Q | 99 |





Table A6.9: Chlorofluorocarbons (C, H, O, N, F, Cl) (... continued)

| Substance Formula (Trivial Name) [CAS Registry Number] InChIKey | $H_s^{cp}$ (at $T^{\ominus}$) $\left[\dfrac{\mathrm{mol}}{\mathrm{m}^3\,\mathrm{Pa}}\right]$ | $\dfrac{\mathrm{d}\ln H_s^{cp}}{\mathrm{d}(1/T)}$ [K] | Reference | Type | Note |
|---|---|---|---|---|---|
| 1-chloro-1,2,2,2-tetrafluoroethane | $1.1\times10^{-4}$ | 2800 | Kutsuna (2013) | M | |
| $C_2HF_4Cl$ | $1.0\times10^{-4}$ | 3500 | Maaßen (1995) | M | 763 |
| (R124) | $1.1\times10^{-4}$ | 3400 | Reichl (1995) | M | 764 |
| [2837-89-0] | $1.1\times10^{-4}$ | | Duchowicz et al. (2020) | V | 186 |
| BOUGCJDAQLKBQH-UHFFFAOYSA-N | $1.0\times10^{-4}$ | 3200 | McLinden (1989) | V | |
| | $1.2\times10^{-4}$ | | Hayer et al. (2022) | Q | 20 |
| | $1.3\times10^{-3}$ | | Duchowicz et al. (2020) | Q | |
| | | 2900 | Kühne et al. (2005) | Q | |
| | | 3400 | Kühne et al. (2005) | ? | |
| 1,2-dichloro-1,1-difluoroethane | $1.4\times10^{-4}$ | | Duchowicz et al. (2020) | V | 186 |
| $C_2H_2Cl_2F_2$ | $1.4\times10^{-4}$ | | HSDB (2015) | V | |
| [1649-08-7] | $1.1\times10^{-3}$ | | Duchowicz et al. (2020) | Q | |
| SKDFWEPBABSFMG-UHFFFAOYSA-N | | | | | |
| 2-chloro-1,1,1-trifluoroethane | $3.7\times10^{-4}$ | 3600 | Maaßen (1995) | M | 765 |
| $C_2H_2F_3Cl$ | $4.1\times10^{-4}$ | 3500 | Reichl (1995) | M | 766 |
| (R133a) | $3.7\times10^{-4}$ | | Hine and Mookerjee (1975) | V | |
| [75-88-7] | $3.7\times10^{-4}$ | | Irmann (1965) | C | |
| CYXIKYKBLDZZNW-UHFFFAOYSA-N | $4.8\times10^{-4}$ | | Hayer et al. (2022) | Q | 20 |
| | $3.7\times10^{-5}$ | | HSDB (2015) | Q | 99 |
| | $3.0\times10^{-4}$ | | Hilal et al. (2008) | Q | |
| | $4.6\times10^{-4}$ | | English and Carroll (2001) | Q | 230, 231 |
| | $3.9\times10^{-4}$ | | Nirmalakhandan and Speece (1988) | Q | |
| | $2.9\times10^{-4}$ | | Irmann (1965) | Q | |
| | $2.7\times10^{-4}$ | | Abraham and Weathersby (1994) | ? | 21 |
| 1,1-dichloro-1-fluoroethane | $2.9\times10^{-4}$ | 2800 | Kutsuna (2013) | M | |
| $CH_3CFCl_2$ | $2.8\times10^{-4}$ | 3700 | Maaßen (1995) | M | 767 |
| (R141b) | $4.5\times10^{-4}$ | | Duchowicz et al. (2020) | V | 186 |
| [1717-00-6] | $4.5\times10^{-4}$ | | HSDB (2015) | V | |
| FRCHKSNAZZFGCA-UHFFFAOYSA-N | $7.7\times10^{-5}$ | 5200 | McLinden (1989) | V | |
| | $2.9\times10^{-4}$ | | Hayer et al. (2022) | Q | 20 |
| | $6.8\times10^{-4}$ | | Duchowicz et al. (2020) | Q | |
| | $2.9\times10^{-4}$ | 3500 | Li et al. (2019) | Q | 1 |
| | | 3300 | Kühne et al. (2005) | Q | |
| | | 3700 | Kühne et al. (2005) | ? | |
| 1-chloro-1,1-difluoroethane | $1.5\times10^{-4}$ | 2600 | Kutsuna (2013) | M | |
| $CH_3CF_2Cl$ | $1.4\times10^{-4}$ | 3200 | Maaßen (1995) | M | 768 |
| (R142b) | $1.4\times10^{-4}$ | 3200 | Reichl (1995) | M | 769 |
| [75-68-3] | $1.5\times10^{-4}$ | 3000 | Chang and Criddle (1995) | M | 770 |
| BHNZEZWIUMJCGF-UHFFFAOYSA-N | $1.4\times10^{-4}$ | 2500 | McLinden (1989) | V | |
| | $1.9\times10^{-4}$ | | Irmann (1965) | C | 294 |
| | $1.6\times10^{-4}$ | | Hayer et al. (2022) | Q | 20 |
| | $1.4\times10^{-4}$ | 3200 | Li et al. (2019) | Q | 1 |
| | $1.5\times10^{-4}$ | | Modarresi et al. (2007) | Q | 67 |
| | $8.4\times10^{-5}$ | | Yaffe et al. (2003) | Q | 248, 249 |
| | $1.5\times10^{-3}$ | | Katritzky et al. (1998) | Q | |



Table A6.9: Chlorofluorocarbons (C, H, O, N, F, Cl) (...continued)

| Substance Formula (Trivial Name) [CAS Registry Number] InChIKey | $H_s^{cp}$ (at $T^\ominus$) $\left[\dfrac{\text{mol}}{\text{m}^3\,\text{Pa}}\right]$ | $\dfrac{\text{d}\ln H_s^{cp}}{\text{d}(1/T)}$ [K] | Reference | Type | Note |
|---|---|---|---|---|---|
| | $1.5\times10^{-4}$ | | Irmann (1965) | Q | |
| 1-chloro-1,2-difluoroethane $C_2H_3ClF_2$ [338-64-7] UOVSDUIHNGNMBZ-UHFFFAOYSA-N | | 2900 3200 | Kühne et al. (2005) Kühne et al. (2005) | Q ? | |
| 1-chloro-1,1,2-trifluoroethane $C_2H_2F_3Cl$ (R133b) [421-04-5] HILNUELUDBMBJQ-UHFFFAOYSA-N | | 2900 3500 | Kühne et al. (2005) Kühne et al. (2005) | Q ? | |
| 2-chloro-1,1-difluoroethene $C_2HClF_2$ (R1122) [359-10-4] HTHNTJCVPNKCPZ-UHFFFAOYSA-N | $1.7\times10^{-4}$ $1.7\times10^{-4}$ $1.9\times10^{-4}$ $1.1\times10^{-4}$ | 3300 3300 2800 3300 | Maaßen (1995) Reichl (1995) Hayer et al. (2022) Kühne et al. (2005) Kühne et al. (2005) Abraham and Weathersby (1994) | M M Q Q ? ? | 771 772 20 21 |
| chlorotrifluoroethene $C_2ClF_3$ [79-38-9] UUAGAQFQZIEFAH-UHFFFAOYSA-N | $3.2\times10^{-5}$ | | HSDB (2015) | Q | 99 |
| 3,3-dichloro-1,1,1,2,2-pentafluoropropane $CF_3CF_2CHCl_2$ (R225ca) [422-56-0] COAUHYBSXMIJDK-UHFFFAOYSA-N | $9.8\times10^{-5}$ $1.5\times10^{-5}$ $8.4\times10^{-4}$ $9.0\times10^{-5}$ $2.0\times10^{-5}$ $3.0\times10^{-4}$ $1.1\times10^{-4}$ $3.9\times10^{-5}$ $2.0\times10^{-5}$ | 3500 | Kutsuna (2013) Keshavarz et al. (2022) Duchowicz et al. (2020) HSDB (2015) Zhang et al. (2010) Zhang et al. (2010) Zhang et al. (2010) Zhang et al. (2010) Duchowicz et al. (2020) | M Q Q Q Q Q Q Q ? | 99 287, 288 287, 289 287, 290 287, 291 185, 21 |
| 1,3-dichloro-1,1,2,2,3-pentafluoro-propane $CClF_2CF_2CHClF$ (R225cb) [507-55-1] UJIGKESMIPTWJH-UHFFFAOYSA-N | $1.1\times10^{-4}$ $3.6\times10^{-6}$ | 3100 | Kutsuna (2013) HSDB (2015) | M Q | 99 |
| 1-chloro-2-fluorobenzene $C_6H_4ClF$ [348-51-6] ZCJAYDKWZAWMPR-UHFFFAOYSA-N | $3.1\times10^{-3}$ | | Ebert et al. (2023) | ? | 316 |



Table A6.9: Chlorofluorocarbons (C, H, O, N, F, Cl) (...continued)

| Substance Formula (Trivial Name) [CAS Registry Number] InChIKey | $H_s^{cp}$ (at $T^\ominus$) $\left[\dfrac{\text{mol}}{\text{m}^3\,\text{Pa}}\right]$ | $\dfrac{\mathrm{d}\ln H_s^{cp}}{\mathrm{d}(1/T)}$ [K] | Reference | Type | Note |
|---|---|---|---|---|---|
| 1-chloro-3-fluorobenzene C$_6$H$_4$ClF [625-98-9] VZHJIJZEOCBKRA-UHFFFAOYSA-N | $1.6\times10^{-3}$ | | Ebert et al. (2023) | ? | 316 |
| 1-chloro-3-(trifluoromethyl)benzene C$_7$H$_4$ClF$_3$ [98-15-7] YTCGOUNVIAWCMG-UHFFFAOYSA-N | $2.9\times10^{-4}$ $2.8\times10^{-3}$ $1.4\times10^{-3}$ $1.4\times10^{-4}$ | | Zhang et al. (2010) Zhang et al. (2010) Zhang et al. (2010) Zhang et al. (2010) | Q Q Q Q | 287, 288 287, 289 287, 290 287, 291 |
| 1-chloro-4-(trifluoromethyl)benzene C$_7$H$_4$ClF$_3$ [98-56-6] QULYNCCPRWKEMF-UHFFFAOYSA-N | $2.8\times10^{-4}$ $2.9\times10^{-4}$ $3.1\times10^{-3}$ $1.5\times10^{-3}$ $1.4\times10^{-4}$ | | HSDB (2015) Zhang et al. (2010) Zhang et al. (2010) Zhang et al. (2010) Zhang et al. (2010) | Q Q Q Q Q | 99 287, 288 287, 289 287, 290 287, 291 |
| 3-chloro-4-fluorobenzotrifluoride C$_7$H$_3$ClF$_4$ [78068-85-6] BKHVEYHSOXVAOP-UHFFFAOYSA-N | $2.4\times10^{-4}$ $2.5\times10^{-3}$ $8.6\times10^{-4}$ $1.1\times10^{-4}$ | | Zhang et al. (2010) Zhang et al. (2010) Zhang et al. (2010) Zhang et al. (2010) | Q Q Q Q | 287, 288 287, 289 287, 290 287, 291 |
| 3,4-dichlorobenzotrifluoride C$_7$H$_3$Cl$_2$F$_3$ [328-84-7] XILPLWOGHPSJBK-UHFFFAOYSA-N | $3.8\times10^{-4}$ $3.9\times10^{-4}$ $5.3\times10^{-3}$ $2.0\times10^{-3}$ $2.3\times10^{-4}$ | | HSDB (2015) Zhang et al. (2010) Zhang et al. (2010) Zhang et al. (2010) Zhang et al. (2010) | Q Q Q Q Q | 99 287, 288 287, 289 287, 290 287, 291 |
| chlorodifluoroethanoic acid CF$_2$ClCOOH (chlorodifluoroacetic acid) [76-04-0] OAWAZQITIZDJRB-UHFFFAOYSA-N | $2.5\times10^{2}$ $2.5\times10^{2}$ $2.5\times10^{2}$ $2.4\times10^{2}$ $5.6\times10^{2}$ $4.4\times10^{1}$ $9.9\times10^{1}$ $4.9\times10^{-1}$ $1.2\times10^{1}$ $2.4\times10^{2}$ | 10000 10000 10000 10000 | Burkholder et al. (2019) Burkholder et al. (2015) Sander et al. (2011) Bowden et al. (1998a) Keshavarz et al. (2022) Duchowicz et al. (2020) Raventos-Duran et al. (2010) Raventos-Duran et al. (2010) Raventos-Duran et al. (2010) Duchowicz et al. (2020) | L L L M Q Q Q Q Q ? | 242, 243 244 245 185, 21 |
| carbonic chloride fluoride COFCl [353-49-1] OXVVNXMNLYYMOL-UHFFFAOYSA-N | $9.9\times10^{-2}$ | | George et al. (1993) | X | 627 |




Table A6.9: Chlorofluorocarbons (C, H, O, N, F, Cl) (...continued)

| Substance Formula (Trivial Name) [CAS Registry Number] InChIKey | $H_s^{cp}$ (at $T^{\ominus}$) $\left[\dfrac{\mathrm{mol}}{\mathrm{m^3\,Pa}}\right]$ | $\dfrac{\mathrm{d}\ln H_s^{cp}}{\mathrm{d}(1/T)}$ [K] | Reference | Type | Note |
|---|---|---|---|---|---|
| trifluoroacetylchloride CF$_3$COCl [354-32-5] PNQBEPDZQUOCNY-UHFFFAOYSA-N | $2.0\times10^{-2}$ $2.7\times10^{-3}$ $2.0\times10^{-2}$ $1.2\times10^{-2}$ $9.3\times10^{-3}$ $2.2\times10^{-2}$ | | Mirabel et al. (1996) De Bruyn et al. (1995a) George et al. (1994b) Keshavarz et al. (2022) Duchowicz et al. (2020) Duchowicz et al. (2020) | M M M Q Q ? | 449 630 299 185, 21 |
| 2,2-dichloro-1,1-difluoro-1-methoxyethane C$_3$H$_4$Cl$_2$F$_2$O (methoxyflurane) [76-38-0] RFKMCNOHBTXSMU-UHFFFAOYSA-N | $2.9\times10^{-3}$ $1.7\times10^{-3}$ $3.0\times10^{-3}$ $1.7\times10^{-3}$ $2.9\times10^{-3}$ $1.8\times10^{-3}$ $2.9\times10^{-2}$ $1.7\times10^{-2}$ $4.1\times10^{-3}$ $1.8\times10^{-2}$ $2.7\times10^{-3}$ $2.7\times10^{-3}$ $1.7\times10^{-3}$ $2.7\times10^{-3}$ | 4100 4300 3600 4800 4000 | Fogg and Sangster (2003) Steward et al. (1973) Allott et al. (1973) Lerman et al. (1983) Smith et al. (1981b) Stoelting and Longshore (1972) Keshavarz et al. (2022) Duchowicz et al. (2020) Hilal et al. (2008) Modarresi et al. (2007) Kühne et al. (2005) Duchowicz et al. (2020) HSDB (2015) Kühne et al. (2005) Abraham and Weathersby (1994) Abraham et al. (1990) | L L L M M M Q Q Q Q Q ? ? ? ? ? | 14 14 773 14 67 185, 21 419 21 |
| 2-(chlorodifluoromethoxy)-1,1,1-trifluoroethane C$_3$H$_2$ClF$_5$O [33018-78-9] HPDPVPUXPBZBOJ-UHFFFAOYSA-N | $1.7\times10^{-5}$ | | Ebert et al. (2023) | ? | 365 |
| 2-(chlorofluoromethoxy)-1,1,1,2-tetrafluoroethane C$_3$H$_2$ClF$_5$O [56885-28-0] GGEBMRHBGQLKGJ-UHFFFAOYSA-N | $2.2\times10^{-4}$ | | Ebert et al. (2023) | ? | 365 |
| 1-chloro-2,2,2-trifluoroethyl difluoromethyl ether C$_3$H$_2$ClF$_5$O (forane; isoflurane) [26675-46-7] PIWKPBJCKXDKJR-UHFFFAOYSA-N | $2.4\times10^{-4}$ $2.4\times10^{-4}$ $2.4\times10^{-4}$ $2.4\times10^{-4}$ $4.8\times10^{-4}$ $5.7\times10^{-4}$ $1.2\times10^{-3}$ $4.2\times10^{-4}$ $1.2\times10^{-3}$ $3.8\times10^{-4}$ | 5300 4400 | Fogg and Sangster (2003) Steward et al. (1973) Allott et al. (1973) Lerman et al. (1983) Smith et al. (1981b) Keshavarz et al. (2022) Duchowicz et al. (2020) Hilal et al. (2008) Modarresi et al. (2007) Kühne et al. (2005) Goss (2005) | L L L M M Q Q Q Q Q Q | 14 14 14 184 67 |



Table A6.9: Chlorofluorocarbons (C, H, O, N, F, Cl) (. . . continued)

| Substance Formula (Trivial Name) [CAS Registry Number] InChIKey | $H_s^{cp}$ (at $T^\ominus$) $\left[\dfrac{\mathrm{mol}}{\mathrm{m^3\,Pa}}\right]$ | $\dfrac{\mathrm{d}\ln H_s^{cp}}{\mathrm{d}(1/T)}$ [K] | Reference | Type | Note |
|---|---|---|---|---|---|
| | $3.4\times10^{-4}$ | | Duchowicz et al. (2020) | ? | 185, 21 |
| | $3.4\times10^{-4}$ | | HSDB (2015) | ? | 419 |
| | | 4500 | Kühne et al. (2005) | ? | |
| | $2.4\times10^{-4}$ | | Abraham and Weathersby (1994) | ? | 21 |
| | $3.4\times10^{-4}$ | | Abraham et al. (1990) | ? | |
| 2-chloro-1,1,2-trifluoroethyl difluoromethyl ether C$_3$H$_2$ClF$_5$O (enflurane) [13838-16-9] JPGQOUSTVILISH-UHFFFAOYSA-N | $3.0\times10^{-4}$ | | Fogg and Sangster (2003) | L | |
| | $3.0\times10^{-4}$ | | Allott et al. (1973) | L | 14 |
| | $2.7\times10^{-4}$ | | Guitart et al. (1989) | M | 14 |
| | $2.9\times10^{-4}$ | | Lerman et al. (1983) | M | 14 |
| | $1.3\times10^{-3}$ | | HSDB (2015) | V | |
| | $3.0\times10^{-4}$ | | Steward et al. (1973) | C | 14 |
| | $6.9\times10^{-4}$ | | Hilal et al. (2008) | Q | |
| | $3.1\times10^{-4}$ | | Abraham and Weathersby (1994) | ? | 21 |
| 3-[2-chloro-4-(trifluoromethyl)phenoxy]benzoic acid C$_{14}$H$_8$ClF$_3$O$_3$ [63734-62-3] ONKRUAQFUNKYAX-UHFFFAOYSA-N | $6.4\times10^{2}$ | | Zhang et al. (2010) | Q | 287, 288 |
| | $3.3\times10^{2}$ | | Zhang et al. (2010) | Q | 287, 289 |
| | $2.1\times10^{5}$ | | Zhang et al. (2010) | Q | 287, 290 |
| | $2.9\times10^{3}$ | | Zhang et al. (2010) | Q | 287, 291 |
| 3-(2-chloro-4-(trifluoromethyl)phenoxy)phenyl acetate C$_{15}$H$_{10}$ClF$_3$O$_3$ [50594-77-9] KNFRYRRZMRETJX-UHFFFAOYSA-N | 1.1 | | Zhang et al. (2010) | Q | 287, 288 |
| | $2.4\times10^{1}$ | | Zhang et al. (2010) | Q | 287, 289 |
| | $2.9\times10^{1}$ | | Zhang et al. (2010) | Q | 287, 290 |
| | 3.6 | | Zhang et al. (2010) | Q | 287, 291 |
| transfluthrin C$_{15}$H$_{12}$Cl$_2$F$_4$O$_2$ [118712-89-3] DDVNRFNDOPPVQJ-HQJQHLMTSA-N | $2.2\times10^{-1}$ | | Ebert et al. (2023) | ? | 318 |
| tefluthrin C$_{17}$H$_{14}$O$_2$ClF$_7$ [79538-32-2] ZFHGXWPMULPQSE-GPCIZFCYSA-N | $6.2\times10^{-3}$ | | HSDB (2015) | V | |
| | $5.0\times10^{-3}$ | | Maniere et al. (2011) | ? | 12, 165 |
| clobetasol C$_{22}$H$_{28}$ClFO$_4$ [25122-41-2] FCSHDIVRCWTZOX-DVTGEIKXSA-N | $6.2\times10^{4}$ | | HSDB (2015) | Q | 99 |
| flufenprox C$_{24}$H$_{22}$ClF$_3$O$_3$ [107713-58-6] RURQAJURNPMSSK-UHFFFAOYSA-N | $2.1\times10^{1}$ | | Ebert et al. (2023) | ? | 318 |



Table A6.9: Chlorofluorocarbons (C, H, O, N, F, Cl) (. . . continued)

| Substance Formula (Trivial Name) [CAS Registry Number] InChIKey | $H_s^{cp}$ (at $T^\ominus$) $\left[\dfrac{\text{mol}}{\text{m}^3\,\text{Pa}}\right]$ | $\dfrac{\text{d}\ln H_s^{cp}}{\text{d}(1/T)}$ [K] | Reference | Type | Note |
|---|---|---|---|---|---|
| EINECS 273-236-7 $C_{28}H_{33}Cl_3F_6O_{11}$ [68954-01-8] KVXVLCBFSKMBAR-UHFFFAOYSA-N | $1.5\times10^{14}$ $6.9\times10^{18}$ $2.3\times10^{12}$ $1.6\times10^{15}$ | | Zhang et al. (2010) Zhang et al. (2010) Zhang et al. (2010) Zhang et al. (2010) | Q Q Q Q | 287, 288 287, 289 287, 290 287, 291 |
| 3,5-dichloro-2,4,6-trifluoropyridine $C_5Cl_2F_3N$ [1737-93-5] PKSORSNCSXBXOT-UHFFFAOYSA-N | $1.6$ $7.7\times10^{-4}$ $1.6\times10^{-3}$ $2.7\times10^{-2}$ | | Zhang et al. (2010) Zhang et al. (2010) Zhang et al. (2010) Zhang et al. (2010) | Q Q Q Q | 287, 288 287, 289 287, 290 287, 291 |
| chlorodifluoronitrooxymethane $CClF_2OONO_2$ [70490-95-8] HWHZTSGXJYKIPK-UHFFFAOYSA-N | $2.9\times10^{-2}$ | 5900 | Kanakidou et al. (1995) | E | 774 |
| 1-chloro-2-nitro-4-(trifluoromethyl)-benzene $C_7H_3ClF_3NO_2$ [121-17-5] TZGFQIXRVUHDLE-UHFFFAOYSA-N | $7.2\times10^{-2}$ $1.2\times10^{-1}$ $1.1\times10^{-1}$ $1.0\times10^{-2}$ | | Zhang et al. (2010) Zhang et al. (2010) Zhang et al. (2010) Zhang et al. (2010) | Q Q Q Q | 287, 288 287, 289 287, 290 287, 291 |
| 2-chloro-1,3-dinitro-5-(trifluoromethyl)-benzene $C_7H_2ClF_3N_2O_4$ [393-75-9] HFHAVERNVFNSHL-UHFFFAOYSA-N | $1.8\times10^{1}$ $2.8$ $1.7\times10^{-1}$ $9.0\times10^{-1}$ | | Zhang et al. (2010) Zhang et al. (2010) Zhang et al. (2010) Zhang et al. (2010) | Q Q Q Q | 287, 288 287, 289 287, 290 287, 291 |
| fluroxypyr $C_7H_5Cl_2FN_2O_3$ [69377-81-7] MEFQWPUMEMWTJP-UHFFFAOYSA-N | $5.7\times10^{5}$ $5.9\times10^{9}$ | | HSDB (2015) Maniere et al. (2011) | V ? | 12, 165 |
| fluoroimide $C_{10}H_4Cl_2FNO_2$ [41205-21-4] IPENDKRRWFURRE-UHFFFAOYSA-N | $7.7\times10^{-1}$ | | Ebert et al. (2023) | ? | 318 |
| norflurazon $C_{12}H_9ClF_3N_3O$ [27314-13-2] NVGOPFQZYCNLDU-UHFFFAOYSA-N | $2.9\times10^{4}$ $2.9\times10^{4}$ $5.5\times10^{3}$ | | Duchowicz et al. (2020) HSDB (2015) Duchowicz et al. (2020) | V V Q | 186 |
| fluchloralin $C_{12}H_{13}ClF_3N_3O_4$ [33245-39-5] MNFMIVVPXOGUMX-UHFFFAOYSA-N | $6.6\times10^{-1}$ $7.4\times10^{-1}$ | | HSDB (2015) Mackay et al. (2006d) | V V | |
| flurochloridone $C_{12}H_{10}Cl_2F_3NO$ [61213-25-0] OQZCSNDVOWYALR-UHFFFAOYSA-N | $2.6\times10^{2}$ $8.2\times10^{-1}$ $4.2\times10^{2}$ | | Duchowicz et al. (2020) Duchowicz et al. (2020) Maniere et al. (2011) | V Q ? | 186 165 |



Table A6.9: Chlorofluorocarbons (C, H, O, N, F, Cl) (. . . continued)

| Substance Formula (Trivial Name) [CAS Registry Number] InChIKey | $H_s^{cp}$ (at $T^{\ominus}$) $\left[\dfrac{\mathrm{mol}}{\mathrm{m^3\,Pa}}\right]$ | $\dfrac{\mathrm{d}\ln H_s^{cp}}{\mathrm{d}(1/T)}$ [K] | Reference | Type | Note |
|---|---|---|---|---|---|
| flupyradifurone $C_{12}H_{11}ClF_2N_2O_2$ [951659-40-8] QOIYTRGFOFZNKF-UHFFFAOYSA-N | $6.5\times10^6$ | | Ebert et al. (2023) | ? | 318 |
| fluxofenim $C_{12}H_{11}NO_3ClF_3$ [88485-37-4] UKSLKNUCVPZQCQ-GZTJUZNOSA-N | 2.6 $5.8\times10^2$ | | Duchowicz et al. (2020) Duchowicz et al. (2020) | V Q | 186 |
| fluazinam $C_{13}H_4Cl_2F_6N_4O_4$ [79622-59-6] UZCGKGPEKUCDTF-UHFFFAOYSA-N | $3.9\times10^{-2}$ $3.9\times10^{-2}$ | | HSDB (2015) Maniere et al. (2011) | V ? | 12, 165 |
| 5-(2-chloro-4-(trifluoromethyl)phenoxy)-2-nitrophenol $C_{13}H_7ClF_3NO_4$ [42874-63-5] WYTRKEWETULQOA-UHFFFAOYSA-N | 9.9 $1.1\times10^3$ $2.3\times10^6$ $2.3\times10^1$ | | Zhang et al. (2010) Zhang et al. (2010) Zhang et al. (2010) Zhang et al. (2010) | Q Q Q Q | 287, 288 287, 289 287, 290 287, 291 |
| florpyrauxifen $C_{13}H_8Cl_2F_2N_2O_3$ [943832-81-3] XFZUQTKDBCOXPP-UHFFFAOYSA-N | $3.2\times10^7$ | | Ebert et al. (2023) | ? | 318 |
| tetraconazole $C_{13}H_{11}Cl_2F_4N_3O$ [112281-77-3] LQDARGUHUSPFNL-UHFFFAOYSA-N | $2.3\times10^3$ $2.8\times10^3$ | | HSDB (2015) Maniere et al. (2011) | V ? | 241, 165 |
| fluopicolide $C_{14}H_8Cl_3F_3N_2O$ [239110-15-7] GBOYJIHYACSLGN-UHFFFAOYSA-N | $9.0\times10^3$ $2.4\times10^4$ | | HSDB (2015) Maniere et al. (2011) | V ? | 241, 165 |
| diflubenzuron $C_{14}H_9ClF_2N_2O_2$ (difluron) [35367-38-5] QQQYTWIFVNKMRW-UHFFFAOYSA-N | $2.1\times10^3$ $2.1\times10^3$ $2.1\times10^3$ $2.1\times10^1$ $7.2\times10^3$ $3.3\times10^1$ $1.7\times10^3$ | | Duchowicz et al. (2020) HSDB (2015) Mackay et al. (2006d) Barcelo and Hennion (1997) Duchowicz et al. (2020) Goodarzi et al. (2010) Maniere et al. (2011) | V V V X Q Q ? | 186 567 568 165 |
| efavirenz $C_{14}H_9ClF_3NO_2$ [154598-52-4] XPOQHMRABVBWPR-ZDUSSCGKSA-N | $1.4\times10^3$ | | HSDB (2015) | Q | 99 |



Table A6.9: Chlorofluorocarbons (C, H, O, N, F, Cl) (... continued)

| Substance<br>Formula<br>(Trivial Name)<br>[CAS Registry Number]<br>InChIKey | $H_s^{cp}$<br>(at $T^{\ominus}$)<br>$\left[\dfrac{\text{mol}}{\text{m}^3\,\text{Pa}}\right]$ | $\dfrac{\mathrm{d}\ln H_s^{cp}}{\mathrm{d}(1/T)}$<br><br>[K] | Reference | Type | Note |
|---|---|---|---|---|---|
| halauxifen-methyl | $8.1\times10^5$ | | Maniere et al. (2011) | ? | 241, 570, 165 |
| $C_{14}H_{11}Cl_2FN_2O_3$ | $8.2\times10^5$ | | Maniere et al. (2011) | ? | 241, 493, 165 |
| [943831-98-9] | $8.3\times10^5$ | | Maniere et al. (2011) | ? | 241, 573, 165 |
| KDHKOPYYWOHESS-UHFFFAOYSA-N | $9.0\times10^5$ | | Maniere et al. (2011) | ? | 241, 165 |
| fluroxypyr-butometyl<br>$C_{14}H_{19}Cl_2FN_2O_4$<br>[154486-27-8]<br>ZKFARSBUEBZZJT-UHFFFAOYSA-N | $2.4\times10^3$ | | Ebert et al. (2023) | ? | 318 |
| teflubenzuron<br>$C_{14}H_6N_2O_2Cl_2F_4$<br>[83121-18-0]<br>CJDWRQLODFKPEL-UHFFFAOYSA-N | $6.2\times10^4$<br>$2.5\times10^4$ | | Duchowicz et al. (2020)<br>Duchowicz et al. (2020) | V<br>Q | 186 |
| fluroxypyr-meptyl<br>$C_{15}H_{21}Cl_2FN_2O_3$<br>(attain A)<br>[81406-37-3]<br>OLZQTUCTGLHFTQ-UHFFFAOYSA-N | $3.7\times10^1$ | | Maniere et al. (2011) | ? | 12, 165 |
| quinoxyfen<br>$C_{15}H_8Cl_2FNO$<br>[124495-18-7]<br>WRPIRSINYZBGPK-UHFFFAOYSA-N | $1.0\times10^3$ | | HSDB (2015) | Q | 99 |
| 5-(2-chloro-4-(trifluoromethyl)phenoxy)-2-nitrophenyl acetate<br>$C_{15}H_9ClF_3NO_5$<br>[50594-44-0]<br>PSWSPFSDVZVVDC-UHFFFAOYSA-N | $2.7\times10^2$<br><br>$1.5\times10^3$<br>$3.7\times10^4$<br>$3.1\times10^2$ | | Zhang et al. (2010)<br><br>Zhang et al. (2010)<br>Zhang et al. (2010)<br>Zhang et al. (2010) | Q<br><br>Q<br>Q<br>Q | 287, 288<br><br>287, 289<br>287, 290<br>287, 291 |
| triflumuron<br>$C_{15}H_{10}ClF_3N_2O_3$<br>[64628-44-0]<br>XAIPTRIXGHTTNT-UHFFFAOYSA-N | $1.1\times10^3$ | | Ebert et al. (2023) | ? | 318 |
| oxyfluorfen<br>$C_{15}H_{11}ClF_3NO_4$<br>[42874-03-3]<br>OQMBBFQZGJFLBU-UHFFFAOYSA-N | $1.2\times10^1$<br>$1.2\times10^1$<br>$8.3\times10^1$<br>$4.2\times10^1$ | | Duchowicz et al. (2020)<br>HSDB (2015)<br>Duchowicz et al. (2020)<br>Maniere et al. (2011) | V<br>V<br>Q<br>? | 186<br><br><br>165 |
| pyraflufen-ethyl<br>$C_{15}H_{13}Cl_2F_3N_2O_4$<br>[129630-19-9]<br>APTZNLHMIGJTEW-UHFFFAOYSA-N | $1.2\times10^4$<br>$4.5\times10^4$ | | MacBean (2012b)<br>Maniere et al. (2011) | X<br>? | 350<br>12, 165 |





Table A6.9: Chlorofluorocarbons (C, H, O, N, F, Cl) (...continued)

| Substance Formula (Trivial Name) [CAS Registry Number] InChIKey | $H_s^{cp}$ (at $T^{\ominus}$) $\left[\dfrac{\mathrm{mol}}{\mathrm{m^3\,Pa}}\right]$ | $\dfrac{\mathrm{d}\ln H_s^{cp}}{\mathrm{d}(1/T)}$ [K] | Reference | Type | Note |
|---|---|---|---|---|---|
| *cis*-furconazole C$_{15}$H$_{14}$Cl$_2$F$_3$N$_3$O$_2$ [112839-32-4] ULCWZQJLFZEXCS-KGLIPLIRSA-N | $3.7\times10^3$ | | Ebert et al. (2023) | ? | 318 |
| carfentrazone ethyl C$_{15}$H$_{14}$Cl$_2$F$_3$N$_3$O$_3$ [128639-02-1] MLKCGVHIFJBRCD-UHFFFAOYSA-N | $3.3\times10^3$ $3.3\times10^3$ $1.7\times10^4$ $6.4\times10^3$ | | Duchowicz et al. (2020) HSDB (2015) Duchowicz et al. (2020) Maniere et al. (2011) | V V Q ? | 186 12, 165 |
| triflumizole C$_{15}$H$_{15}$ClF$_3$N$_3$O [68694-11-1] HSMVPDGQOIQYSR-UHFFFAOYSA-N | $1.9\times10^5$ $2.5\times10^7$ $3.7\times10^1$ | | Duchowicz et al. (2020) Mackay et al. (2006d) Duchowicz et al. (2020) | V V Q | 186 |
| diflumetorim C$_{15}$H$_{16}$N$_3$OClF$_2$ [130339-07-0] NEKULYKCZPJMMJ-UHFFFAOYSA-N | $3.1\times10^2$ $2.8\times10^3$ | | Duchowicz et al. (2020) Duchowicz et al. (2020) | V Q | 186 |
| 3-(2,4-dichlorophenyl)-6-fluoro-2-(1H-1,2,4-triazol-1-yl)-quinazolin-4(3H)-one C$_{16}$H$_8$Cl$_2$FN$_5$O (fluquinconazole) [136426-54-5] IJJVMEJXYNJXOJ-UHFFFAOYSA-N | $5.6\times10^8$ $1.4\times10^5$ $4.8\times10^5$ | | Hilal et al. (2008) Modarresi et al. (2007) Maniere et al. (2011) | Q Q ? | 67 241, 165 |
| hexaflumuron C$_{16}$H$_8$Cl$_2$F$_6$N$_2$O$_3$ [86479-06-3] RGNPBRKPHBKNKX-UHFFFAOYSA-N | $9.9\times10^{-1}$ $9.9\times10^{-1}$ $4.3\times10^4$ | | Duchowicz et al. (2020) HSDB (2015) Duchowicz et al. (2020) | V V Q | 186 |
| fluopyram C$_{16}$H$_{11}$ClF$_6$N$_2$O [658066-35-4] KVDJTXBXMWJJEF-UHFFFAOYSA-N | $3.4\times10^4$ | | Maniere et al. (2011) | ? | 241, 165 |
| haloxyfop-p-methyl C$_{16}$H$_{13}$ClF$_3$NO$_4$ [72619-32-0] MFSWTRQUCLNFOM-SECBINFHSA-N | $8.3\times10^2$ | | Maniere et al. (2011) | ? | 241, 165 |
| flumetralin C$_{16}$H$_{12}$ClF$_4$N$_3$O$_4$ [62924-70-3] PWNAWOCHVWERAR-UHFFFAOYSA-N | 5.2 $4.0\times10^3$ | | Duchowicz et al. (2020) Duchowicz et al. (2020) | V Q | 186 |



Table A6.9: Chlorofluorocarbons (C, H, O, N, F, Cl) (. . . continued)

| Substance Formula (Trivial Name) [CAS Registry Number] InChIKey | $H_s^{cp}$ (at $T^{\ominus}$) $\left[\dfrac{\text{mol}}{\text{m}^3\,\text{Pa}}\right]$ | $\dfrac{\text{d}\ln H_s^{cp}}{\text{d}(1/T)}$ [K] | Reference | Type | Note |
|---|---|---|---|---|---|
| haloxyfop-methyl $C_{16}H_{13}ClF_3NO_4$ [69806-40-2] MFSWTRQUCLNFOM-UHFFFAOYSA-N | $3.1\times10^1$ $3.1\times10^2$ | | Duchowicz et al. (2020) Duchowicz et al. (2020) | V Q | 186 |
| pydiflumetofen $C_{16}H_{16}Cl_3F_2N_3O_2$ [1228284-64-7] DGOAXBPOVUPPEB-UHFFFAOYSA-N | $6.6\times10^3$ | | Ebert et al. (2023) | ? | 318 |
| nuarimol $C_{17}H_{12}ClFN_2O$ [63284-71-9] SAPGTCDSBGMXCD-UHFFFAOYSA-N | $1.5\times10^7$ | | MacBean (2012a) | ? | |
| clodinafop-propargyl $C_{17}H_{13}ClFNO_4$ [105512-06-9] JBDHZKLJNAIJNC-LLVKDONJSA-N | $3.6\times10^3$ $3.5\times10^3$ $2.4\times10^3$ $3.6\times10^3$ | | Duchowicz et al. (2020) HSDB (2015) Duchowicz et al. (2020) Maniere et al. (2011) | V V Q ? | 186 241, 165 |
| flamprop-methyl $C_{17}H_{15}ClFNO_3$ [52756-25-9] RBNIGDFIUWJJEV-UHFFFAOYSA-N | $2.2\times10^3$ | | MacBean (2012a) | ? | |
| pyridalyl $C_{18}H_{14}Cl_4F_3NO_3$ [179101-81-6] UBSUICNYBBPFHD-UHFFFAOYSA-N | $4.9\times10^6$ | | HSDB (2015) | V | |
| mefentrifluconazole $C_{18}H_{15}ClF_3N_3O_2$ [1417782-03-6] JERZEQUMJNCPRJ-UHFFFAOYSA-N | $6.2\times10^2$ | | Maniere et al. (2011) | ? | 12, 165 |
| bixafen $C_{18}H_{12}N_3OCl_2F_3$ [581809-46-3] LDLMOOXUCMHBMZ-UHFFFAOYSA-N | $2.6\times10^4$ | | Maniere et al. (2011) | ? | 241, 165 |
| benzovindiflupyr $C_{18}H_{15}Cl_2F_2N_3O$ [1072957-71-1] CCCGEKHKTPTUHJ-UHFFFAOYSA-N | $7.7\times10^5$ | | Maniere et al. (2011) | ? | 241, 165 |
| lactofen $C_{19}H_{15}ClF_3NO_7$ [77501-63-4] CONWAEURSVPLRM-UHFFFAOYSA-N | $2.1\times10^1$ $2.3\times10^1$ $4.0\times10^4$ | | Duchowicz et al. (2020) HSDB (2015) Duchowicz et al. (2020) | V V Q | 186 |



Table A6.9: Chlorofluorocarbons (C, H, O, N, F, Cl) (...continued)

| Substance<br>Formula<br>(Trivial Name)<br>[CAS Registry Number]<br>InChIKey | $H_s^{cp}$<br>(at $T^\ominus$)<br>$\left[\dfrac{\mathrm{mol}}{\mathrm{m^3\,Pa}}\right]$ | $\dfrac{\mathrm{d}\ln H_s^{cp}}{\mathrm{d}(1/T)}$<br><br>[K] | Reference | Type | Note |
|---|---|---|---|---|---|
| flupoxam<br>$C_{19}H_{14}N_4O_2ClF_5$<br>[119126-15-7]<br>AOQMRUTZEYVDIL-UHFFFAOYSA-N | $1.1\times10^2$<br>$2.7\times10^7$ | | Duchowicz et al. (2020)<br>Duchowicz et al. (2020) | V<br>Q | 186 |
| flamprop-m-isopropyl<br>$C_{19}H_{19}NO_3ClF$<br>[63782-90-1]<br>IKVXBIIHQGXQRQ-CYBMUJFWSA-N | $3.9\times10^2$<br>$2.3\times10^1$ | | Duchowicz et al. (2020)<br>Duchowicz et al. (2020) | V<br>Q | 186 |
| haloxyfop-ethoxyethyl<br>$C_{19}H_{19}ClF_3NO_5$<br>[87237-48-7]<br>MIJLZGZLQLAQCM-UHFFFAOYSA-N | $3.8\times10^4$ | | Ebert et al. (2023) | ? | 318 |
| chlorfluazuron<br>$C_{20}H_9Cl_3F_5N_3O_3$<br>[71422-67-8]<br>UISUNVFOGSJSKD-UHFFFAOYSA-N | $8.2\times10^6$<br>$1.1\times10^6$<br>$3.9\times10^8$<br>$2.1\times10^6$<br>$5.5\times10^6$ | | Keshavarz et al. (2022)<br>Duchowicz et al. (2020)<br>Hilal et al. (2008)<br>Modarresi et al. (2007)<br>Duchowicz et al. (2020) | Q<br>Q<br>Q<br>Q<br>? | <br><br><br>67<br>185, 21 |
| fluazuron<br>$C_{20}H_{10}N_3O_3Cl_2F_5$<br>[86811-58-7]<br>YOWNVPAUWYHLQX-UHFFFAOYSA-N | $>2.3\times10^{10}$ | | MacBean (2012a) | ? | |
| flufenoxuron<br>$C_{21}H_{11}ClF_6N_2O_3$<br>[101463-69-8]<br>RYLHNOVXKPXDIP-UHFFFAOYSA-N | $3.8\times10^6$ | | HSDB (2015) | Q | 99 |
| fluoxastrobin<br>$C_{21}H_{16}N_4O_5ClF$<br>[361377-29-9]<br>UFEODZBUAFNAEU-NLRVBDNBSA-N | $9.0\times10^6$<br>$1.0\times10^7$ | | HSDB (2015)<br>Maniere et al. (2011) | V<br>? | <br>12, 165 |
| haloperidol<br>$C_{21}H_{23}ClFNO_2$<br>[52-86-8]<br>LNEPOXFFQSENCJ-UHFFFAOYSA-N | $4.3\times10^8$ | | HSDB (2015) | Q | 99 |
| indoxacarb<br>$C_{22}H_{17}ClF_3N_3O_7$<br>[173584-44-6]<br>VBCVPMMZEGZULK-NRFANRHFSA-N | $1.5\times10^4$<br>$>1.7\times10^4$ | | HSDB (2015)<br>Maniere et al. (2011) | V<br>? | <br>165 |
| cyfluthrin<br>$C_{22}H_{18}Cl_2FNO_3$<br>[68359-37-5]<br>QQODLKZGRKWIFG-UHFFFAOYSA-N | $3.4\times10^2$<br>$1.1\times10^1$<br>$1.7$ | | HSDB (2015)<br>Maniere et al. (2011)<br>Maniere et al. (2011) | V<br>?<br>? | <br>12, 165<br>12, 165 |



Table A6.9: Chlorofluorocarbons (C, H, O, N, F, Cl) (...continued)

| Substance Formula (Trivial Name) [CAS Registry Number] InChIKey | $H_s^{cp}$ (at $T^\ominus$) $\left[\dfrac{\text{mol}}{\text{m}^3\,\text{Pa}}\right]$ | $\dfrac{\mathrm{d}\ln H_s^{cp}}{\mathrm{d}(1/T)}$ [K] | Reference | Type | Note |
|---|---|---|---|---|---|
| cyfluthrin I $C_{22}H_{18}Cl_2FNO_3$ [86560-92-1] QQODLKZGRKWIFG-QSFXBCCZSA-N | 2.2 | | Ebert et al. (2023) | ? | 318 |
| $\beta$-cyfluthrin (cis) $C_{22}H_{18}Cl_2FNO_3$ [86560-93-2] QQODLKZGRKWIFG-XFQXTVEOSA-N | $1.4\times10^2$ | | Ebert et al. (2023) | ? | 365 |
| cyfluthrin II $C_{22}H_{18}Cl_2FNO_3$ [86560-94-3] QQODLKZGRKWIFG-BPAFIMBUSA-N | $1.1\times10^2$ | | Ebert et al. (2023) | ? | 365 |
| $\beta$-cyfluthrin (trans) $C_{22}H_{18}Cl_2FNO_3$ [86560-95-4] QQODLKZGRKWIFG-QKYXUNIQSA-N | $3.6\times10^1$ | | Ebert et al. (2023) | ? | 365 |
| metamifop $C_{23}H_{18}N_2O_4ClF$ [256412-89-2] ADDQHLREJDZPMT-CQSZACIVSA-N | $1.6\times10^1$ | | MacBean (2012a) | ? | 12 |
| pyriminostrobin $C_{23}H_{18}Cl_2F_3N_3O_4$ [1257598-43-8] YYXSCUSVVALMNW-FOWTUZBSSA-N | $8.3\times10^3$ | | Ebert et al. (2023) | ? | 318 |
| cyhalothrin $C_{23}H_{19}NO_3ClF_3$ [68085-85-8] ZXQYGBMAQZUVMI-BWHPXCRDSA-N | $7.0\times10^{-1}$ | | HSDB (2015) | Q | 99 |
| bifenthrin $C_{23}H_{22}ClF_3O_2$ [82657-04-3] OMFRMAHOUUJSGP-UNOMPAQXSA-N | 9.9 9.9 9.9 $2.7\times10^{-1}$ 4.7 | | Duchowicz et al. (2020) HSDB (2015) Hilal et al. (2008) Duchowicz et al. (2020) Hilal et al. (2008) | V V C Q Q | 186 |
| $\lambda$-cyhalothrin $C_{23}H_{19}ClF_3NO_3$ [91465-08-6] DFVKOWFGNASVPK-QQDHXZELSA-N | $5.0\times10^1$ | | Maniere et al. (2011) | ? | 12, 165 |
| $\gamma$-cyhalothrin $C_{23}H_{19}ClF_3NO_3$ [76703-62-3] ZXQYGBMAQZUVMI-GCMPRSNUSA-N | $4.5\times10^1$ | | Maniere et al. (2011) | ? | 241, 165 |





Table A6.9: Chlorofluorocarbons (C, H, O, N, F, Cl) (...continued)

| Substance Formula (Trivial Name) [CAS Registry Number] InChIKey | $H_s^{cp}$ (at $T^{\ominus}$) $\left[\dfrac{\mathrm{mol}}{\mathrm{m^3\,Pa}}\right]$ | $\dfrac{\mathrm{d}\ln H_s^{cp}}{\mathrm{d}(1/T)}$ [K] | Reference | Type | Note |
|---|---|---|---|---|---|
| flucycloxuron $C_{25}H_{20}ClF_2N_3O_3$ [94050-52-9] PCKNFPQPGUWFHO-UQRQXUALSA-N | $3.8\times10^1$ | | MacBean (2012a) | ? | |
| $\tau$-fluvalinate $C_{26}H_{22}ClF_3N_2O_3$ [102851-06-9] INISTDXBRIBGOC-XMMISQBUSA-N | $8.3\times10^3$ | | Maniere et al. (2011) | ? | 165 |
| fluvalinate $C_{26}H_{22}ClF_3N_2O_3$ [69409-94-5] INISTDXBRIBGOC-UHFFFAOYSA-N | $6.6\times10^2$ | | HSDB (2015) | Q | 99 |



## A7 Organic species with bromine (Br)

### A7.1 Bromocarbons (C, H, O, N, Br)

Table A7.1: Bromocarbons (C, H, O, N, Br)

| Substance Formula (Trivial Name) [CAS Registry Number] InChIKey | $H_s^{cp}$ (at $T^\ominus$) $\left[\dfrac{\mathrm{mol}}{\mathrm{m}^3\,\mathrm{Pa}}\right]$ | $\dfrac{\mathrm{d}\ln H_s^{cp}}{\mathrm{d}(1/T)}$ [K] | Reference | Type | Note |
|---|---|---|---|---|---|
| bromomethane | $1.5\times10^{-3}$ | 3300 | Burkholder et al. (2019) | L | 775 |
| $CH_3Br$ | $1.5\times10^{-3}$ | 3000 | Burkholder et al. (2019) | L | 70 |
| (methyl bromide) | $1.7\times10^{-3}$ | 3100 | Burkholder et al. (2015) | L | |
| [74-83-9] | $1.5\times10^{-3}$ | 3000 | Burkholder et al. (2015) | L | 70 |
| GZUXJHMPEANEGY-UHFFFAOYSA-N | $1.5\times10^{-3}$ | 3800 | Brockbank (2013) | L | 1 |
| | $1.7\times10^{-3}$ | 3100 | Sander et al. (2011) | L | |
| | $1.7\times10^{-3}$ | 3100 | Sander et al. (2006) | L | |
| | $1.7\times10^{-3}$ | 3100 | Staudinger and Roberts (2001) | L | |
| | $1.6\times10^{-3}$ | 3100 | Wilhelm et al. (1977) | L | |
| | $1.3\times10^{-3}$ | 2800 | Hiatt (2013) | M | |
| | $2.0\times10^{-3}$ | | Thomas et al. (2006) | M | 154, 703 |
| | $1.8\times10^{-3}$ | 2500 | De Bruyn and Saltzman (1997) | M | 776 |
| | $1.4\times10^{-3}$ | | Gan and Yates (1996) | M | 294 |
| | $1.7\times10^{-3}$ | 3400 | Elliott and Rowland (1993) | M | |
| | $1.5\times10^{-3}$ | 2600 | Swain and Thornton (1962) | M | |
| | $1.6\times10^{-3}$ | 3200 | Glew and Moelwyn-Hughes (1953) | M | 777 |
| | $1.6\times10^{-3}$ | | Mackay et al. (2006b) | V | |
| | $1.6\times10^{-3}$ | | Lide and Frederikse (1995) | V | |
| | $1.6\times10^{-3}$ | | Mackay et al. (1993) | V | |
| | $1.9\times10^{-3}$ | | Mackay and Shiu (1981) | V | 12 |
| | $1.5\times10^{-3}$ | | Hine and Mookerjee (1975) | V | |
| | $1.5\times10^{-3}$ | | Yaws (2003) | X | 237 |
| | $4.4\times10^{-5}$ | 350 | Goldstein (1982) | X | 298 |
| | $1.8\times10^{-4}$ | | Keshavarz et al. (2022) | Q | |
| | $3.5\times10^{-3}$ | | Duchowicz et al. (2020) | Q | 184 |
| | $5.5\times10^{-4}$ | | Wang et al. (2017) | Q | 80, 238 |
| | $3.8\times10^{-3}$ | | Wang et al. (2017) | Q | 80, 239 |
| | $2.3\times10^{-3}$ | | Wang et al. (2017) | Q | 80, 240 |
| | $3.7\times10^{-3}$ | | Gharagheizi et al. (2012) | Q | |
| | $1.2\times10^{-3}$ | | Raventos-Duran et al. (2010) | Q | 242, 243 |
| | $3.1\times10^{-3}$ | | Raventos-Duran et al. (2010) | Q | 244 |
| | $1.2\times10^{-3}$ | | Raventos-Duran et al. (2010) | Q | 245 |
| | $1.7\times10^{-3}$ | | Gharagheizi et al. (2010) | Q | 246 |
| | $2.3\times10^{-3}$ | | Modarresi et al. (2007) | Q | 67 |
| | | 3400 | Kühne et al. (2005) | Q | |
| | $1.9\times10^{-3}$ | | Yaffe et al. (2003) | Q | 248, 249 |
| | $1.8\times10^{-3}$ | | Yao et al. (2002) | Q | 229 |
| | $1.2\times10^{-3}$ | | English and Carroll (2001) | Q | 230, 231 |
| | $9.9\times10^{-5}$ | | Katritzky et al. (1998) | Q | |
| | $1.5\times10^{-3}$ | | Suzuki et al. (1992) | Q | 232 |
| | $3.1\times10^{-3}$ | | Nirmalakhandan and Speece (1988) | Q | |
| | $7.9\times10^{-4}$ | | Irmann (1965) | Q | |
| | $1.3\times10^{-3}$ | | Duchowicz et al. (2020) | ? | 185, 21 |



Table A7.1: Bromocarbons (C, H, O, N, Br) (. . . continued)

| Substance Formula (Trivial Name) [CAS Registry Number] InChIKey | $H_s^{cp}$ (at $T^\ominus$) $\left[\dfrac{\mathrm{mol}}{\mathrm{m^3\,Pa}}\right]$ | $\dfrac{\mathrm{d}\ln H_s^{cp}}{\mathrm{d}(1/T)}$ [K] | Reference | Type | Note |
|---|---|---|---|---|---|
| | $1.8\times10^{-3}$ | | Thomas et al. (2006) | ? | 154, 704 |
| | | 3200 | Kühne et al. (2005) | ? | |
| | $1.5\times10^{-3}$ | | Yaws (1999) | ? | 21 |
| | $1.7\times10^{-3}$ | | Yates and Gan (1998) | ? | |
| | $1.4\times10^{-3}$ | | Yaws and Yang (1992) | ? | 21 |
| | $1.6\times10^{-3}$ | | Abraham et al. (1990) | ? | |
| dibromomethane | $1.2\times10^{-2}$ | 4300 | Burkholder et al. (2019) | L | 1 |
| $CH_2Br_2$ | $9.0\times10^{-3}$ | 4600 | Burkholder et al. (2019) | L | 70 |
| [74-95-3] | $9.0\times10^{-3}$ | 4600 | Burkholder et al. (2015) | L | 70 |
| FJBFPHVGVWTDIP-UHFFFAOYSA-N | $1.2\times10^{-2}$ | 4300 | Brockbank (2013) | L | 1, 778 |
| | $3.1\times10^{-2}$ | | Mackay and Shiu (1981) | L | |
| | $1.2\times10^{-2}$ | 5000 | Hiatt (2013) | M | |
| | $8.9\times10^{-3}$ | 4400 | Ooki and Yokouchi (2011) | M | 70 |
| | $1.4\times10^{-2}$ | | Dohnal and Hovorka (1999) | M | 12 |
| | $1.5\times10^{-2}$ | | Hovorka and Dohnal (1997) | M | 12 |
| | $1.2\times10^{-2}$ | 4900 | Kondoh and Nakajima (1997) | M | |
| | $9.7\times10^{-3}$ | 3800 | Moore et al. (1995) | M | 779, 70 |
| | $1.1\times10^{-2}$ | 4000 | Wright et al. (1992) | M | 780 |
| | $1.1\times10^{-2}$ | 4100 | Tse et al. (1992) | M | |
| | $1.1\times10^{-2}$ | 4400 | Rex (1906) | M | |
| | $1.1\times10^{-2}$ | | Mackay et al. (2006b) | V | |
| | $1.3\times10^{-2}$ | 4200 | Fogg and Sangster (2003) | V | |
| | $7.1\times10^{-3}$ | | Mackay et al. (1993) | V | |
| | $1.1\times10^{-2}$ | | Hine and Mookerjee (1975) | V | |
| | $1.1\times10^{-2}$ | | Yaws (2003) | X | 237 |
| | $3.2\times10^{-2}$ | | Gharagheizi et al. (2012) | Q | |
| | $6.2\times10^{-3}$ | | Raventos-Duran et al. (2010) | Q | 242, 243 |
| | $4.9\times10^{-2}$ | | Raventos-Duran et al. (2010) | Q | 244 |
| | $9.9\times10^{-3}$ | | Raventos-Duran et al. (2010) | Q | 245 |
| | $1.2\times10^{-1}$ | | Gharagheizi et al. (2010) | Q | 246 |
| | $3.8\times10^{-2}$ | | Hilal et al. (2008) | Q | |
| | $1.7\times10^{-3}$ | | Modarresi et al. (2007) | Q | 67 |
| | | 4500 | Kühne et al. (2005) | Q | |
| | $1.2\times10^{-2}$ | | Yaffe et al. (2003) | Q | 248, 249 |
| | $8.2\times10^{-3}$ | | Yao et al. (2002) | Q | 229, 267 |
| | $2.7\times10^{-3}$ | | Katritzky et al. (1998) | Q | |
| | $9.5\times10^{-3}$ | | Nirmalakhandan and Speece (1988) | Q | |
| | $1.2\times10^{-2}$ | | Mackay et al. (2006b) | ? | |
| | | 4300 | Kühne et al. (2005) | ? | |
| | $1.1\times10^{-2}$ | | Yaws (1999) | ? | 21 |
| | $1.2\times10^{-2}$ | | Mackay et al. (1993) | ? | |
| | $1.1\times10^{-2}$ | | Abraham et al. (1990) | ? | |





Table A7.1: Bromocarbons (C, H, O, N, Br) (... continued)

| Substance Formula (Trivial Name) [CAS Registry Number] InChIKey | $H_s^{cp}$ (at $T^{\ominus}$) $\left[ \dfrac{\text{mol}}{\text{m}^3\,\text{Pa}} \right]$ | $\dfrac{\text{d}\ln H_s^{cp}}{\text{d}(1/T)}$ [K] | Reference | Type | Note |
|---|---|---|---|---|---|
| tribromomethane | $1.7\times10^{-2}$ | 5200 | Burkholder et al. (2019) | L | |
| CHBr$_3$ | $1.1\times10^{-2}$ | 6200 | Burkholder et al. (2019) | L | 70 |
| (bromoform) | $1.7\times10^{-2}$ | 5200 | Burkholder et al. (2015) | L | |
| [75-25-2] | $1.1\times10^{-2}$ | 6200 | Burkholder et al. (2015) | L | 70 |
| DIKBFYAXUHHXCS-UHFFFAOYSA-N | $1.8\times10^{-2}$ | 4800 | Brockbank (2013) | L | 1 |
| | $1.7\times10^{-2}$ | 5200 | Sander et al. (2011) | L | |
| | $1.7\times10^{-2}$ | 5200 | Sander et al. (2006) | L | |
| | $1.7\times10^{-2}$ | 5200 | Staudinger and Roberts (2001) | L | |
| | $1.7\times10^{-2}$ | 5200 | Staudinger and Roberts (1996) | L | |
| | $1.6\times10^{-2}$ | | Mackay and Shiu (1981) | L | |
| | $2.2\times10^{-2}$ | 6300 | Hiatt (2013) | M | |
| | $9.9\times10^{-3}$ | 6200 | Ooki and Yokouchi (2011) | M | 70 |
| | $2.0\times10^{-2}$ | | Ruiz-Bevia and Fernandez-Torres (2010) | M | |
| | $9.6\times10^{-3}$ | | Zhang et al. (2002) | M | 14 |
| | $2.3\times10^{-2}$ | | Hovorka and Dohnal (1997) | M | 12 |
| | $1.4\times10^{-2}$ | 4500 | Kondoh and Nakajima (1997) | M | |
| | $1.5\times10^{-2}$ | 4300 | Moore et al. (1995) | M | 781, 70 |
| | $8.5\times10^{-3}$ | 1500 | Khalfaoui and Newsham (1994a) | M | |
| | $2.4\times10^{-2}$ | 4100 | Wright et al. (1992) | M | 782 |
| | $1.9\times10^{-2}$ | 5000 | Tse et al. (1992) | M | |
| | $1.8\times10^{-2}$ | 4700 | Munz and Roberts (1987) | M | |
| | $1.6\times10^{-2}$ | 5700 | Nicholson et al. (1984) | M | |
| | $1.9\times10^{-2}$ | | Warner et al. (1980) | M | |
| | $1.7\times10^{-2}$ | | Mackay et al. (2006b) | V | |
| | $1.8\times10^{-2}$ | 5300 | Fogg and Sangster (2003) | V | |
| | $1.7\times10^{-2}$ | | Mackay et al. (1993) | V | |
| | $1.7\times10^{-2}$ | | Warner et al. (1980) | V | |
| | $1.5\times10^{-2}$ | | Hine and Mookerjee (1975) | V | |
| | $1.8\times10^{-2}$ | 2700 | Goldstein (1982) | X | 298 |
| | $1.7\times10^{-2}$ | | Ryan et al. (1988) | C | |
| | $1.7\times10^{-2}$ | | Nicholson et al. (1984) | C | |
| | $1.9\times10^{-2}$ | | Shen (1982) | C | |
| | $8.8\times10^{-3}$ | | Keshavarz et al. (2022) | Q | |
| | $1.0\times10^{-2}$ | | Duchowicz et al. (2020) | Q | |
| | $7.1\times10^{-2}$ | | Gharagheizi et al. (2012) | Q | |
| | $4.9\times10^{-3}$ | | Raventos-Duran et al. (2010) | Q | 242, 243 |
| | $6.2\times10^{-3}$ | | Raventos-Duran et al. (2010) | Q | 244 |
| | $7.8\times10^{-2}$ | | Raventos-Duran et al. (2010) | Q | 245 |
| | $7.3\times10^{-3}$ | | Hilal et al. (2008) | Q | |
| | $9.8\times10^{-4}$ | | Modarresi et al. (2007) | Q | 67 |
| | | 5600 | Kühne et al. (2005) | Q | |
| | $1.8\times10^{-2}$ | | Yaffe et al. (2003) | Q | 248, 249 |
| | $2.3\times10^{-2}$ | | Yao et al. (2002) | Q | 229 |
| | $1.4\times10^{-2}$ | | Katritzky et al. (1998) | Q | |
| | $2.4\times10^{-2}$ | | Nirmalakhandan and Speece (1988) | Q | |
| | $1.8\times10^{-2}$ | | Duchowicz et al. (2020) | ? | 185, 21 |





Table A7.1: Bromocarbons (C, H, O, N, Br) (. . . continued)

| Substance<br>Formula<br>(Trivial Name)<br>[CAS Registry Number]<br>InChIKey | $H_s^{cp}$<br>(at $T^\ominus$)<br>$\left[\dfrac{\text{mol}}{\text{m}^3\,\text{Pa}}\right]$ | $\dfrac{\text{d}\ln H_s^{cp}}{\text{d}(1/T)}$<br><br>[K] | Reference | Type | Note |
|---|---|---|---|---|---|
| | $2.1\times10^{-2}$ | | Mackay et al. (2006b) | ? | |
| | | 5000 | Kühne et al. (2005) | ? | |
| | $1.7\times10^{-2}$ | | Yaws (1999) | ? | 21 |
| | $2.1\times10^{-2}$ | | Mackay et al. (1993) | ? | |
| | $1.5\times10^{-2}$ | | Abraham et al. (1990) | ? | |
| tetrabromomethane | $2.0\times10^{-2}$ | | Duchowicz et al. (2020) | V | 186 |
| CBr$_4$ | $2.0\times10^{-2}$ | | HSDB (2015) | V | |
| [558-13-4] | $1.2\times10^{-2}$ | | Fogg and Sangster (2003) | V | 783, 38 |
| HJUGFYREWKUQJT-UHFFFAOYSA-N | $2.0\times10^{-2}$ | | Hilal et al. (2008) | C | |
| | $9.8\times10^{-4}$ | | Duchowicz et al. (2020) | Q | |
| | $2.1\times10^{-3}$ | | Hilal et al. (2008) | Q | |
| | $2.2\times10^{-4}$ | | Modarresi et al. (2007) | Q | 67 |
| bromoethane | $1.3\times10^{-3}$ | 3900 | Burkholder et al. (2019) | L | |
| C$_2$H$_5$Br | $1.3\times10^{-3}$ | 3900 | Brockbank (2013) | L | 1 |
| [74-96-4] | $1.3\times10^{-3}$ | | Li et al. (1993) | M | |
| RDHPKYGYEGBMSE-UHFFFAOYSA-N | $1.3\times10^{-3}$ | 3900 | Rex (1906) | M | |
| | $1.3\times10^{-3}$ | | Duchowicz et al. (2020) | V | 186 |
| | | | Mackay et al. (2006b) | V | 683 |
| | $8.1\times10^{-4}$ | | Mackay et al. (1993) | V | |
| | $1.4\times10^{-3}$ | | Abraham (1984) | V | |
| | $1.3\times10^{-3}$ | | Hine and Mookerjee (1975) | V | |
| | $1.3\times10^{-3}$ | | Yaws (2003) | X | 237 |
| | $9.2\times10^{-5}$ | | Ryan et al. (1988) | C | |
| | $1.2\times10^{-3}$ | | Duchowicz et al. (2020) | Q | |
| | $3.2\times10^{-3}$ | | Gharagheizi et al. (2012) | Q | |
| | $9.9\times10^{-4}$ | | Raventos-Duran et al. (2010) | Q | 271, 243 |
| | $3.1\times10^{-3}$ | | Raventos-Duran et al. (2010) | Q | 244 |
| | $9.9\times10^{-4}$ | | Raventos-Duran et al. (2010) | Q | 245 |
| | $1.4\times10^{-3}$ | | Gharagheizi et al. (2010) | Q | 246 |
| | $3.4\times10^{-3}$ | | Hilal et al. (2008) | Q | |
| | $1.2\times10^{-3}$ | | Modarresi et al. (2007) | Q | 67 |
| | | 3700 | Kühne et al. (2005) | Q | |
| | $1.4\times10^{-3}$ | | Yaffe et al. (2003) | Q | 248, 249 |
| | $7.7\times10^{-4}$ | | Yao et al. (2002) | Q | 229 |
| | $1.1\times10^{-3}$ | | English and Carroll (2001) | Q | 230, 231 |
| | $2.3\times10^{-4}$ | | Katritzky et al. (1998) | Q | |
| | $1.1\times10^{-3}$ | | Suzuki et al. (1992) | Q | 232 |
| | $1.6\times10^{-3}$ | | Nirmalakhandan and Speece (1988) | Q | |
| | | 3800 | Kühne et al. (2005) | ? | |
| | $1.3\times10^{-3}$ | | Yaws (1999) | ? | 21 |
| | $1.3\times10^{-3}$ | | Yaws and Yang (1992) | ? | 21 |
| | $1.4\times10^{-3}$ | | Abraham et al. (1990) | ? | |



Table A7.1: Bromocarbons (C, H, O, N, Br) (... continued)

| Substance Formula (Trivial Name) [CAS Registry Number] InChIKey | $H_s^{cp}$ (at $T^\ominus$) $\left[\dfrac{\mathrm{mol}}{\mathrm{m^3\,Pa}}\right]$ | $\dfrac{\mathrm{d\ln} H_s^{cp}}{\mathrm{d}(1/T)}$ [K] | Reference | Type | Note |
|---|---|---|---|---|---|
| 1,1-dibromoethane $C_2H_4Br_2$ [557-91-5] APQIUTYORBAGEZ-UHFFFAOYSA-N | $7.6\times10^{-3}$ | | HSDB (2015) | Q | 99 |
| 1,2-dibromoethane $C_2H_4Br_2$ (ethylene dibromide) [106-93-4] PAAZPARNPHGIKF-UHFFFAOYSA-N | $1.4\times10^{-2}$ | 4300 | Burkholder et al. (2019) | L | |
| | $1.5\times10^{-2}$ | 3900 | Burkholder et al. (2015) | L | |
| | $1.4\times10^{-2}$ | 4200 | Brockbank (2013) | L | 1 |
| | $1.7\times10^{-2}$ | 5500 | Hiatt (2013) | M | |
| | $1.9\times10^{-2}$ | | Dohnal and Hovorka (1999) | M | 12 |
| | $1.3\times10^{-2}$ | | Welke et al. (1998) | M | |
| | $1.9\times10^{-2}$ | | Hovorka and Dohnal (1997) | M | 12 |
| | $1.8\times10^{-2}$ | 5500 | Kondoh and Nakajima (1997) | M | |
| | $1.1\times10^{-2}$ | 3000 | Khalfaoui and Newsham (1994a) | M | 33 |
| | $1.5\times10^{-2}$ | 3900 | Ashworth et al. (1988) | M | 278 |
| | $1.5\times10^{-2}$ | | Mackay et al. (2006b) | V | |
| | $2.1\times10^{-3}$ | | Mackay et al. (1993) | V | |
| | $1.4\times10^{-2}$ | | Hine and Mookerjee (1975) | V | |
| | $1.2\times10^{-2}$ | | Yaws (2003) | X | 237 |
| | $1.1\times10^{-2}$ | 1900 | Goldstein (1982) | X | 298 |
| | $1.5\times10^{-2}$ | | HSDB (2015) | C | |
| | $1.2\times10^{-2}$ | | Keshavarz et al. (2022) | Q | |
| | $3.3\times10^{-3}$ | | Duchowicz et al. (2020) | Q | 184 |
| | $6.2\times10^{-3}$ | | Wang et al. (2017) | Q | 80, 238 |
| | $4.6\times10^{-2}$ | | Wang et al. (2017) | Q | 80, 239 |
| | $4.4\times10^{-2}$ | | Wang et al. (2017) | Q | 80, 240 |
| | $2.8\times10^{-2}$ | | Gharagheizi et al. (2012) | Q | |
| | $1.2\times10^{-2}$ | | Gharagheizi et al. (2010) | Q | 246 |
| | $3.9\times10^{-2}$ | | Hilal et al. (2008) | Q | |
| | $6.5\times10^{-3}$ | | Modarresi et al. (2007) | Q | 67 |
| | | 4800 | Kühne et al. (2005) | Q | |
| | $1.6\times10^{-2}$ | | Yaffe et al. (2003) | Q | 248, 249 |
| | $4.1\times10^{-3}$ | | Katritzky et al. (1998) | Q | |
| | $7.5\times10^{-3}$ | | Nirmalakhandan and Speece (1988) | Q | |
| | $1.2\times10^{-3}$ | | Rumble (2021) | ? | 784, 785 |
| | $1.5\times10^{-2}$ | | Duchowicz et al. (2020) | ? | 185, 21 |
| | $1.5\times10^{-2}$ | | Mackay et al. (2006b) | ? | |
| | | 4200 | Kühne et al. (2005) | ? | |
| | $1.3\times10^{-2}$ | | Yaws (1999) | ? | 21 |
| | $1.5\times10^{-2}$ | | Mackay et al. (1993) | ? | |
| | $1.4\times10^{-2}$ | | Yaws and Yang (1992) | ? | 21 |
| | $2.1\times10^{-2}$ | | Abraham et al. (1990) | ? | |
| | $1.6\times10^{-2}$ | | Mackay and Yeun (1983) | ? | |
| | $1.8\times10^{-2}$ | | Chiou et al. (1980) | ? | 79 |



Table A7.1: Bromocarbons (C, H, O, N, Br) (...continued)

| Substance Formula (Trivial Name) [CAS Registry Number] InChIKey | $H_s^{cp}$ (at $T^\ominus$) $\left[\dfrac{\mathrm{mol}}{\mathrm{m^3\,Pa}}\right]$ | $\dfrac{\mathrm{d}\ln H_s^{cp}}{\mathrm{d}(1/T)}$ [K] | Reference | Type | Note |
|---|---|---|---|---|---|
| 1,2-dibromoethane-d4 $C_2D_4Br_2$ (ethylene dibromide-d4) [22581-63-1] PAAZPARNPHGIKF-LNLMKGTHSA-N | $1.6\times10^{-2}$ | 4800 | Hiatt (2013) | M | |
| 1,1,2-tribromoethane $C_2H_3Br_3$ [78-74-0] QUMDOMSJJIFTCA-UHFFFAOYSA-N | $2.4\times10^{-2}$ | | Ebert et al. (2023) | ? | 316 |
| 1,1,2,2-tetrabromoethane $C_2H_2Br_4$ [79-27-6] QXSZNDIIPUOQMB-UHFFFAOYSA-N | $1.4\times10^{-2}$ | 1300 | Burkholder et al. (2019) | L | |
| | $1.4\times10^{-2}$ | 1300 | Brockbank (2013) | L | 1 |
| | $1.0\times10^{-2}$ | 840 | Khalfaoui and Newsham (1994a) | M | |
| | $7.4\times10^{-1}$ | | Duchowicz et al. (2020) | V | 186 |
| | $7.6\times10^{-1}$ | | HSDB (2015) | V | |
| | $7.3\times10^{-1}$ | | Yaws (2003) | X | 237 |
| | $1.4\times10^{-2}$ | | Duchowicz et al. (2020) | Q | |
| | $5.7\times10^{-1}$ | | Zhang et al. (2010) | Q | 287, 288 |
| | $2.9\times10^{-1}$ | | Zhang et al. (2010) | Q | 287, 289 |
| | $4.3\times10^{-1}$ | | Zhang et al. (2010) | Q | 287, 290 |
| | $1.5\times10^{-1}$ | | Zhang et al. (2010) | Q | 287, 291 |
| | $7.2\times10^{-1}$ | | Gharagheizi et al. (2010) | Q | 246 |
| | $2.4\times10^{-1}$ | | Hilal et al. (2008) | Q | |
| | $7.0\times10^{-1}$ | | Yaffe et al. (2003) | Q | 248, 249 |
| | $2.3\times10^{-1}$ | | Katritzky et al. (1998) | Q | |
| | $7.3\times10^{-1}$ | | Yaws (1999) | ? | 21 |
| 1-bromopropane $C_3H_7Br$ [106-94-5] CYNYIHKIEHGYOZ-UHFFFAOYSA-N | $1.1\times10^{-3}$ | 4600 | Brockbank (2013) | L | 1 |
| | $1.1\times10^{-3}$ | | Li et al. (1993) | M | |
| | $1.1\times10^{-3}$ | 4500 | Rex (1906) | M | |
| | $1.4\times10^{-3}$ | | HSDB (2015) | V | |
| | $2.6\times10^{-4}$ | | Mackay et al. (2006b) | V | |
| | $2.6\times10^{-4}$ | | Mackay et al. (1993) | V | |
| | $1.0\times10^{-3}$ | | Abraham (1984) | V | |
| | $1.0\times10^{-3}$ | | Hine and Mookerjee (1975) | V | |
| | $1.3\times10^{-3}$ | | Yaws (2003) | X | 237, 12 |
| | $1.5\times10^{-3}$ | | Gharagheizi et al. (2012) | Q | |
| | $6.2\times10^{-4}$ | | Raventos-Duran et al. (2010) | Q | 271, 243 |
| | $2.0\times10^{-3}$ | | Raventos-Duran et al. (2010) | Q | 244 |
| | $6.2\times10^{-4}$ | | Raventos-Duran et al. (2010) | Q | 245 |
| | $1.1\times10^{-3}$ | | Gharagheizi et al. (2010) | Q | 246 |
| | $2.8\times10^{-3}$ | | Hilal et al. (2008) | Q | |
| | $1.2\times10^{-3}$ | | Modarresi et al. (2007) | Q | 67 |
| | $1.4\times10^{-3}$ | | Yaffe et al. (2003) | Q | 248, 249 |
| | $3.9\times10^{-4}$ | | Yao et al. (2002) | Q | 229 |
| | $8.4\times10^{-4}$ | | English and Carroll (2001) | Q | 230, 274 |
| | $3.4\times10^{-4}$ | | Katritzky et al. (1998) | Q | |
| | $3.5\times10^{-4}$ | | Russell et al. (1992) | Q | 279 |



Table A7.1: Bromocarbons (C, H, O, N, Br) (...continued)

| Substance Formula (Trivial Name) [CAS Registry Number] InChIKey | $H_s^{cp}$ (at $T^\ominus$) $\left[\dfrac{\text{mol}}{\text{m}^3\,\text{Pa}}\right]$ | $\dfrac{\text{d}\ln H_s^{cp}}{\text{d}(1/T)}$ [K] | Reference | Type | Note |
|---|---|---|---|---|---|
|  | $8.4\times10^{-4}$ |  | Suzuki et al. (1992) | Q | 232 |
|  | $1.3\times10^{-3}$ |  | Nirmalakhandan and Speece (1988) | Q |  |
|  | $1.1\times10^{-3}$ |  | Yaws (1999) | ? | 21, 12 |
|  | $1.4\times10^{-3}$ |  | Yaws and Yang (1992) | ? | 21, 12 |
|  | $1.0\times10^{-3}$ |  | Abraham et al. (1990) | ? |  |
| 2-bromopropane $C_3H_7Br$ [75-26-3] NAMYKGVDVNBCFQ-UHFFFAOYSA-N | $9.0\times10^{-4}$ | 3900 | Brockbank (2013) | L | 1 |
|  | $8.4\times10^{-4}$ |  | Li et al. (1993) | M |  |
|  | $9.0\times10^{-4}$ | 4500 | Rex (1906) | M |  |
|  | $9.0\times10^{-4}$ |  | Duchowicz et al. (2020) | V | 186 |
|  | $9.0\times10^{-4}$ |  | HSDB (2015) | V |  |
|  | $7.9\times10^{-4}$ |  | Mackay et al. (2006b) | V |  |
|  | $7.9\times10^{-4}$ |  | Mackay et al. (1993) | V |  |
|  | $9.0\times10^{-4}$ |  | Hine and Mookerjee (1975) | V |  |
|  | $1.0\times10^{-3}$ |  | Yaws (2003) | X | 237, 12 |
|  | $5.4\times10^{-4}$ |  | Duchowicz et al. (2020) | Q |  |
|  | $3.6\times10^{-3}$ |  | Gharagheizi et al. (2012) | Q |  |
|  | $6.2\times10^{-4}$ |  | Raventos-Duran et al. (2010) | Q | 242, 243 |
|  | $1.2\times10^{-3}$ |  | Raventos-Duran et al. (2010) | Q | 244 |
|  | $6.2\times10^{-4}$ |  | Raventos-Duran et al. (2010) | Q | 245 |
|  | $9.4\times10^{-4}$ |  | Gharagheizi et al. (2010) | Q | 246 |
|  | $1.5\times10^{-3}$ |  | Hilal et al. (2008) | Q |  |
|  | $7.5\times10^{-4}$ |  | Modarresi et al. (2007) | Q | 67 |
|  | $9.2\times10^{-4}$ |  | Yaffe et al. (2003) | Q | 248, 249 |
|  | $3.4\times10^{-4}$ |  | Yao et al. (2002) | Q | 229, 267 |
|  | $6.7\times10^{-4}$ |  | English and Carroll (2001) | Q | 230, 231 |
|  | $3.3\times10^{-4}$ |  | Katritzky et al. (1998) | Q |  |
|  | $7.3\times10^{-4}$ |  | Suzuki et al. (1992) | Q | 232 |
|  | $9.2\times10^{-4}$ |  | Nirmalakhandan and Speece (1988) | Q |  |
|  | $8.1\times10^{-4}$ |  | Yaws (1999) | ? | 21, 12 |
|  | $1.0\times10^{-3}$ |  | Yaws and Yang (1992) | ? | 21, 12 |
|  | $9.0\times10^{-4}$ |  | Abraham et al. (1990) | ? |  |
| 1,2-dibromopropane $C_3H_6Br_2$ [78-75-1] XFNJYAKDBJUJAJ-UHFFFAOYSA-N | $5.3\times10^{-3}$ |  | Albanese et al. (1987) | M |  |
|  | $6.8\times10^{-3}$ |  | Duchowicz et al. (2020) | V | 186 |
|  | $6.8\times10^{-3}$ |  | HSDB (2015) | V |  |
|  | $6.8\times10^{-3}$ |  | Mackay et al. (2006b) | V |  |
|  | $1.1\times10^{-2}$ |  | Hine and Mookerjee (1975) | V |  |
|  | $6.3\times10^{-3}$ |  | Yaws (2003) | X | 237 |
|  | $1.5\times10^{-3}$ |  | Duchowicz et al. (2020) | Q |  |
|  | $3.5\times10^{-2}$ |  | Gharagheizi et al. (2012) | Q |  |
|  | $6.6\times10^{-3}$ |  | Gharagheizi et al. (2010) | Q | 246 |
|  | $1.9\times10^{-2}$ |  | Hilal et al. (2008) | Q |  |
|  | $2.8\times10^{-3}$ |  | Modarresi et al. (2007) | Q | 67 |
|  | $6.9\times10^{-3}$ |  | Yaffe et al. (2003) | Q | 248, 249 |
|  | $5.2\times10^{-3}$ |  | Katritzky et al. (1998) | Q |  |
|  | $4.4\times10^{-3}$ |  | Nirmalakhandan and Speece (1988) | Q |  |
|  | $6.6\times10^{-3}$ |  | Yaws and Yang (1992) | ? | 21 |



Table A7.1: Bromocarbons (C, H, O, N, Br) (...continued)

| Substance<br>Formula<br>(Trivial Name)<br>[CAS Registry Number]<br>InChIKey | $H_s^{cp}$<br>(at $T^\ominus$)<br>$\left[\dfrac{\text{mol}}{\text{m}^3\,\text{Pa}}\right]$ | $\dfrac{\text{d}\ln H_s^{cp}}{\text{d}(1/T)}$<br><br>[K] | Reference | Type | Note |
|---|---|---|---|---|---|
| 1,3-dibromopropane | $1.1\times10^{-3}$ | | Mackay et al. (1993) | V | |
| $C_3H_6Br_2$ | $1.1\times10^{-2}$ | | Hine and Mookerjee (1975) | V | |
| [109-64-8] | $1.6\times10^{-2}$ | | Keshavarz et al. (2022) | Q | |
| VEFLKXRACNJHOV-UHFFFAOYSA-N | $3.0\times10^{-3}$ | | Duchowicz et al. (2020) | Q | 299 |
| | $7.2\times10^{-2}$ | | Hilal et al. (2008) | Q | |
| | $5.0\times10^{-3}$ | | Modarresi et al. (2007) | Q | 67 |
| | $1.2\times10^{-2}$ | | Yaffe et al. (2003) | Q | 248, 249 |
| | $5.6\times10^{-3}$ | | Katritzky et al. (1998) | Q | |
| | $6.0\times10^{-3}$ | | Nirmalakhandan and Speece (1988) | Q | |
| | $1.1\times10^{-2}$ | | Duchowicz et al. (2020) | ? | 185, 21 |
| 1-bromobutane | $7.4\times10^{-4}$ | | Brockbank (2013) | L | |
| $C_4H_9Br$ | $4.6\times10^{-4}$ | | Hoff et al. (1993) | M | |
| [109-65-9] | $8.2\times10^{-4}$ | | Li et al. (1993) | M | |
| MPPPKRYCTPRNTB-UHFFFAOYSA-N | $1.1\times10^{-3}$ | | Duchowicz et al. (2020) | V | 186 |
| | $1.1\times10^{-3}$ | | HSDB (2015) | V | |
| | $8.0\times10^{-4}$ | | Abraham (1984) | V | |
| | $8.0\times10^{-4}$ | | Hine and Mookerjee (1975) | V | |
| | $8.0\times10^{-4}$ | | Yaws (2003) | X | 237 |
| | $1.2\times10^{-3}$ | | Duchowicz et al. (2020) | Q | |
| | $9.6\times10^{-4}$ | | Gharagheizi et al. (2012) | Q | |
| | $4.9\times10^{-4}$ | | Raventos-Duran et al. (2010) | Q | 242, 243 |
| | $1.6\times10^{-3}$ | | Raventos-Duran et al. (2010) | Q | 244 |
| | $4.9\times10^{-4}$ | | Raventos-Duran et al. (2010) | Q | 245 |
| | $7.5\times10^{-4}$ | | Gharagheizi et al. (2010) | Q | 246 |
| | $2.2\times10^{-3}$ | | Hilal et al. (2008) | Q | |
| | $8.7\times10^{-4}$ | | Modarresi et al. (2007) | Q | 67 |
| | $1.1\times10^{-3}$ | | Yaffe et al. (2003) | Q | 248, 249 |
| | $2.7\times10^{-4}$ | | Yao et al. (2002) | Q | 229 |
| | $6.2\times10^{-4}$ | | English and Carroll (2001) | Q | 230, 231 |
| | $4.1\times10^{-4}$ | | Katritzky et al. (1998) | Q | |
| | $5.8\times10^{-4}$ | | Russell et al. (1992) | Q | 279 |
| | $6.5\times10^{-4}$ | | Suzuki et al. (1992) | Q | 232 |
| | $1.0\times10^{-3}$ | | Nirmalakhandan and Speece (1988) | Q | |
| | $8.3\times10^{-4}$ | | Haynes (2014) | ? | 786 |
| | $8.1\times10^{-4}$ | | Yaws (1999) | ? | 21 |
| | $8.1\times10^{-4}$ | | Yaws and Yang (1992) | ? | 21 |
| | $7.9\times10^{-4}$ | | Abraham et al. (1990) | ? | |
| 2-bromobutane | $8.8\times10^{-4}$ | | Brockbank (2013) | L | |
| $C_4H_9Br$ | $7.7\times10^{-4}$ | | Li et al. (1993) | M | |
| [78-76-2] | $6.2\times10^{-4}$ | | HSDB (2015) | Q | 99 |
| UPSXAPQYNGXVBF-UHFFFAOYSA-N | $1.4\times10^{-3}$ | | Hilal et al. (2008) | Q | |



Table A7.1: Bromocarbons (C, H, O, N, Br) (... continued)

| Substance<br>Formula<br>(Trivial Name)<br>[CAS Registry Number]<br>InChIKey | $H_s^{cp}$<br>(at $T^\ominus$)<br>$\left[\dfrac{\mathrm{mol}}{\mathrm{m^3\,Pa}}\right]$ | $\dfrac{\mathrm{d}\ln H_s^{cp}}{\mathrm{d}(1/T)}$<br><br>[K] | Reference | Type | Note |
|---|---|---|---|---|---|
| 1-bromo-2-methylpropane | $4.2\times10^{-4}$ | | Hine and Mookerjee (1975) | V | |
| $C_4H_9Br$ | $4.9\times10^{-4}$ | | Raventos-Duran et al. (2010) | Q | 242, 243 |
| [78-77-3] | $1.6\times10^{-3}$ | | Raventos-Duran et al. (2010) | Q | 244 |
| HLVFKOKELQSXIQ-UHFFFAOYSA-N | $4.9\times10^{-4}$ | | Raventos-Duran et al. (2010) | Q | 245 |
| | $2.0\times10^{-3}$ | | Hilal et al. (2008) | Q | |
| | $7.7\times10^{-4}$ | | Modarresi et al. (2007) | Q | 67 |
| | $4.2\times10^{-4}$ | | Yaffe et al. (2003) | Q | 248, 249 |
| | $4.0\times10^{-4}$ | | English and Carroll (2001) | Q | 230, 231 |
| | $8.6\times10^{-4}$ | | Nirmalakhandan et al. (1997) | Q | |
| | $5.8\times10^{-4}$ | | Suzuki et al. (1992) | Q | 232 |
| | $4.2\times10^{-4}$ | | Abraham et al. (1990) | ? | |
| 2-bromo-2-methylpropane | $2.4\times10^{-4}$ | | Duchowicz et al. (2020) | V | 186 |
| $C_4H_9Br$ | $2.4\times10^{-4}$ | | HSDB (2015) | V | |
| [507-19-7] | $2.0\times10^{-4}$ | | Duchowicz et al. (2020) | Q | |
| RKSOPLXZQNSWAS-UHFFFAOYSA-N | $4.9\times10^{-4}$ | | Raventos-Duran et al. (2010) | Q | 242, 243 |
| | $6.2\times10^{-4}$ | | Raventos-Duran et al. (2010) | Q | 244 |
| | $4.9\times10^{-4}$ | | Raventos-Duran et al. (2010) | Q | 245 |
| | $5.2\times10^{-4}$ | | Hilal et al. (2008) | Q | |
| | $5.1\times10^{-4}$ | | Modarresi et al. (2007) | Q | 67 |
| | $4.5\times10^{-4}$ | | English and Carroll (2001) | Q | 230, 231 |
| | $4.2\times10^{-4}$ | | Katritzky et al. (1998) | Q | |
| | $5.2\times10^{-4}$ | | Nirmalakhandan et al. (1997) | Q | |
| | $3.1\times10^{-4}$ | | Yaws and Yang (1992) | ? | 21, 28 |
| | $9.7\times10^{-5}$ | | Abraham et al. (1990) | ? | |
| 1-bromo-3-methylbutane | $4.9\times10^{-4}$ | | Mackay et al. (1993) | V | |
| $C_5H_{11}Br$ | $2.9\times10^{-4}$ | | Hine and Mookerjee (1975) | V | |
| [107-82-4] | $5.9\times10^{-4}$ | | Keshavarz et al. (2022) | Q | |
| YXZFFTJAHVMMLF-UHFFFAOYSA-N | $4.5\times10^{-4}$ | | Duchowicz et al. (2020) | Q | 184 |
| | $1.8\times10^{-3}$ | | Hilal et al. (2008) | Q | |
| | $7.8\times10^{-4}$ | | Modarresi et al. (2007) | Q | 67 |
| | $4.7\times10^{-4}$ | | Katritzky et al. (1998) | Q | |
| | $7.0\times10^{-4}$ | | Nirmalakhandan et al. (1997) | Q | |
| | $4.5\times10^{-4}$ | | Suzuki et al. (1992) | Q | 232 |
| | $2.9\times10^{-4}$ | | Duchowicz et al. (2020) | ? | 185, 21 |
| 1,4-dibromobutane | $1.7\times10^{-2}$ | | Albanese et al. (1987) | M | |
| $C_4H_8Br_2$ | $7.3\times10^{-2}$ | | Hilal et al. (2008) | Q | |
| [110-52-1] | $4.3\times10^{-3}$ | | Modarresi et al. (2007) | Q | 67 |
| ULTHEAFYOOPTTB-UHFFFAOYSA-N | | | | | |
| 1-bromopentane | $5.0\times10^{-4}$ | | Duchowicz et al. (2020) | V | 186 |
| $C_5H_{11}Br$ | $4.7\times10^{-4}$ | | Abraham (1984) | V | |
| [110-53-2] | $5.6\times10^{-4}$ | | Yaws (2003) | X | 237 |
| YZWKKMVJZFACSU-UHFFFAOYSA-N | $1.2\times10^{-3}$ | | Duchowicz et al. (2020) | Q | |
| | $6.1\times10^{-4}$ | | Gharagheizi et al. (2012) | Q | |
| | $5.1\times10^{-4}$ | | Gharagheizi et al. (2010) | Q | 246 |
| | $1.8\times10^{-3}$ | | Hilal et al. (2008) | Q | |



Table A7.1: Bromocarbons (C, H, O, N, Br) (...continued)

| Substance Formula (Trivial Name) [CAS Registry Number] InChIKey | $H_s^{cp}$ (at $T^\ominus$) $\left[\dfrac{\text{mol}}{\text{m}^3\,\text{Pa}}\right]$ | $\dfrac{\text{d}\ln H_s^{cp}}{\text{d}(1/T)}$ [K] | Reference | Type | Note |
|---|---|---|---|---|---|
| | $6.5\times10^{-4}$ | | Modarresi et al. (2007) | Q | 67 |
| | $5.1\times10^{-4}$ | | Yaffe et al. (2003) | Q | 248, 249 |
| | $4.6\times10^{-4}$ | | English and Carroll (2001) | Q | 230, 231 |
| | $5.1\times10^{-4}$ | | Katritzky et al. (1998) | Q | |
| | $8.0\times10^{-4}$ | | Nirmalakhandan et al. (1997) | Q | |
| | $5.6\times10^{-4}$ | | Yaws (1999) | ? | 21 |
| | $5.0\times10^{-4}$ | | Yaws and Yang (1992) | ? | 21 |
| | $4.7\times10^{-4}$ | | Abraham et al. (1990) | ? | |
| 1-bromo-2-methylbutane C$_5$H$_{11}$Br [10422-35-2] XKVLZBNEPALHIO-UHFFFAOYSA-N | $8.8\times10^{-4}$ | | Nirmalakhandan and Speece (1988) | Q | |
| 1-bromohexane C$_6$H$_{13}$Br [111-25-1] MNDIARAMWBIKFW-UHFFFAOYSA-N | $3.0\times10^{-4}$ | | Duchowicz et al. (2020) | V | 186 |
| | $3.0\times10^{-4}$ | | Abraham (1984) | V | |
| | $1.2\times10^{-3}$ | | Duchowicz et al. (2020) | Q | |
| | $3.1\times10^{-4}$ | | Raventos-Duran et al. (2010) | Q | 242, 243 |
| | $7.8\times10^{-4}$ | | Raventos-Duran et al. (2010) | Q | 244 |
| | $3.1\times10^{-4}$ | | Raventos-Duran et al. (2010) | Q | 245 |
| | $1.5\times10^{-3}$ | | Hilal et al. (2008) | Q | |
| | $5.2\times10^{-4}$ | | Modarresi et al. (2007) | Q | 67 |
| | $2.2\times10^{-4}$ | | Yaffe et al. (2003) | Q | 248, 272 |
| | $3.4\times10^{-4}$ | | English and Carroll (2001) | Q | 230, 231 |
| | $5.3\times10^{-4}$ | | Katritzky et al. (1998) | Q | |
| | $6.2\times10^{-4}$ | | Nirmalakhandan et al. (1997) | Q | |
| | $3.0\times10^{-4}$ | | Abraham et al. (1990) | ? | |
| 1-bromo-3-methylpentane C$_6$H$_{13}$Br [51116-73-5] MDCCBJLCTOTLKM-UHFFFAOYSA-N | $2.3\times10^{-4}$ | | English and Carroll (2001) | Q | 230, 231 |
| | $5.8\times10^{-4}$ | | Nirmalakhandan and Speece (1988) | Q | |
| bromocyclohexane C$_6$H$_{11}$Br [108-85-0] AQNQQHJNRPDOQV-UHFFFAOYSA-N | $7.0\times10^{-3}$ | | Hilal et al. (2008) | Q | |
| 1-bromoheptane C$_7$H$_{15}$Br [629-04-9] LSXKDWGTSHCFPP-UHFFFAOYSA-N | $2.2\times10^{-4}$ | | Brockbank (2013) | L | |
| | $2.2\times10^{-4}$ | | Duchowicz et al. (2020) | V | 186 |
| | $2.3\times10^{-4}$ | | Abraham (1984) | V | |
| | $2.2\times10^{-4}$ | | Yaws (2003) | X | 237 |
| | $1.1\times10^{-3}$ | | Duchowicz et al. (2020) | Q | |
| | $3.5\times10^{-4}$ | | Gharagheizi et al. (2012) | Q | |
| | $2.0\times10^{-4}$ | | Raventos-Duran et al. (2010) | Q | 242, 243 |
| | $6.2\times10^{-4}$ | | Raventos-Duran et al. (2010) | Q | 244 |
| | $2.0\times10^{-4}$ | | Raventos-Duran et al. (2010) | Q | 245 |
| | $2.3\times10^{-4}$ | | Gharagheizi et al. (2010) | Q | 246 |
| | $1.2\times10^{-3}$ | | Hilal et al. (2008) | Q | |
| | $4.7\times10^{-4}$ | | Modarresi et al. (2007) | Q | 67 |



Table A7.1: Bromocarbons (C, H, O, N, Br) (... continued)

| Substance Formula (Trivial Name) [CAS Registry Number] InChIKey | $H_s^{cp}$ (at $T^{\ominus}$) $\left[\dfrac{\mathrm{mol}}{\mathrm{m^3\,Pa}}\right]$ | $\dfrac{\mathrm{d}\ln H_s^{cp}}{\mathrm{d}(1/T)}$ [K] | Reference | Type | Note |
|---|---|---|---|---|---|
| | $2.2\times10^{-4}$ | | Yaffe et al. (2003) | Q | 248, 249 |
| | $6.5\times10^{-5}$ | | Yao et al. (2002) | Q | 229 |
| | $2.6\times10^{-4}$ | | English and Carroll (2001) | Q | 230, 274 |
| | $5.6\times10^{-4}$ | | Katritzky et al. (1998) | Q | |
| | $5.0\times10^{-4}$ | | Nirmalakhandan et al. (1997) | Q | |
| | $2.2\times10^{-4}$ | | Yaws (1999) | ? | 21 |
| | $2.3\times10^{-4}$ | | Abraham et al. (1990) | ? | |
| 1-bromooctane $C_8H_{17}Br$ [111-83-1] VMKOFRJSULQZRM-UHFFFAOYSA-N | $1.7\times10^{-4}$ | | Duchowicz et al. (2020) | V | 186 |
| | $2.4\times10^{-4}$ | 4600 | Sarraute et al. (2004) | V | |
| | $1.7\times10^{-4}$ | | Abraham (1984) | V | |
| | $1.1\times10^{-3}$ | | Duchowicz et al. (2020) | Q | |
| | $1.6\times10^{-4}$ | | Raventos-Duran et al. (2010) | Q | 271, 243 |
| | $4.9\times10^{-4}$ | | Raventos-Duran et al. (2010) | Q | 244 |
| | $1.6\times10^{-4}$ | | Raventos-Duran et al. (2010) | Q | 245 |
| | $9.7\times10^{-4}$ | | Hilal et al. (2008) | Q | |
| | $4.0\times10^{-4}$ | | Modarresi et al. (2007) | Q | 67 |
| | $1.7\times10^{-4}$ | | Yaffe et al. (2003) | Q | 248, 249 |
| | $1.9\times10^{-4}$ | | English and Carroll (2001) | Q | 230, 231 |
| | $5.6\times10^{-4}$ | | Katritzky et al. (1998) | Q | |
| | $3.9\times10^{-4}$ | | Nirmalakhandan et al. (1997) | Q | |
| | $1.7\times10^{-4}$ | | Abraham et al. (1990) | ? | |
| 1,8-dibromooctane $C_8H_{16}Br_2$ [4549-32-0] DKEGCUDAFWNSSO-UHFFFAOYSA-N | $1.3\times10^{-2}$ | 6900 | Sarraute et al. (2006) | M | 787 |
| 1,2-dibromo-4-(1,2-dibromoethyl)cyclohexane $C_8H_{12}Br_4$ [3322-93-8] PQRRSJBLKOPVJV-UHFFFAOYSA-N | $1.7\times10^{2}$ | | HSDB (2015) | Q | 99 |
| | $2.4\times10^{-1}$ | | Zhang et al. (2010) | Q | 287, 288 |
| | $2.9$ | | Zhang et al. (2010) | Q | 287, 289 |
| | $1.0\times10^{1}$ | | Zhang et al. (2010) | Q | 287, 290 |
| | $2.5\times10^{-1}$ | | Zhang et al. (2010) | Q | 287, 291 |
| 1-bromononane $C_9H_{19}Br$ [693-58-3] AYMUQTNXKPEMLM-UHFFFAOYSA-N | $7.9\times10^{-4}$ | | Hilal et al. (2008) | Q | |
| 1-bromodecane $C_{10}H_{21}Br$ [112-29-8] MYMSJFSOOQERIO-UHFFFAOYSA-N | $1.7\times10^{-4}$ | | Ebert et al. (2023) | ? | 316 |





Table A7.1: Bromocarbons (C, H, O, N, Br) (...continued)

| Substance<br>Formula<br>(Trivial Name)<br>[CAS Registry Number]<br>InChIKey | $H_s^{cp}$<br>(at $T^\ominus$)<br>$\left[\dfrac{\text{mol}}{\text{m}^3\,\text{Pa}}\right]$ | $\dfrac{\text{d}\ln H_s^{cp}}{\text{d}(1/T)}$<br><br>[K] | Reference | Type | Note |
|---|---|---|---|---|---|
| hexabromocyclododecane<br>$C_{12}H_{18}Br_6$<br>[3194-55-6]<br>DEIGXXQKDWULML-UHFFFAOYSA-N | $2.1\times10^{-1}$<br>$2.1\times10^{-1}$<br>$1.6\times10^{-2}$<br>$1.3\times10^{-2}$<br>$5.7$<br>$1.7\times10^{2}$<br>$5.7\times10^{3}$<br>$6.5$ | | Duchowicz et al. (2020)<br>HSDB (2015)<br>HSDB (2015)<br>Duchowicz et al. (2020)<br>Zhang et al. (2010)<br>Zhang et al. (2010)<br>Zhang et al. (2010)<br>Zhang et al. (2010) | V<br>V<br>V<br>Q<br>Q<br>Q<br>Q<br>Q | 186<br><br><br><br>287, 288<br>287, 289<br>287, 290<br>287, 291 |
| vinyl bromide<br>$C_2H_3Br$<br>[593-60-2]<br>INLLPKCGLOXCIV-UHFFFAOYSA-N | $7.0\times10^{-4}$<br>$8.0\times10^{-4}$<br>$7.7\times10^{-4}$<br>$4.8\times10^{-4}$<br>$8.2\times10^{-4}$ | | HSDB (2015)<br>Zhang et al. (2010)<br>Zhang et al. (2010)<br>Zhang et al. (2010)<br>Zhang et al. (2010) | Q<br>Q<br>Q<br>Q<br>Q | 99<br>287, 288<br>287, 289<br>287, 290<br>287, 291 |
| 1,2-dibromoethene<br>$C_2H_2Br_2$<br>[540-49-8]<br>UWTUEMKLYAGTNQ-UHFFFAOYSA-N | $1.2\times10^{-2}$<br>$1.2\times10^{-2}$<br>$4.8\times10^{-3}$ | | Duchowicz et al. (2020)<br>HSDB (2015)<br>Duchowicz et al. (2020) | V<br>V<br>Q | 186 |
| (Z)-1,2-dibromoethene<br>$C_2H_2Br_2$<br>(cis-1,2-dibromoethene)<br>[590-11-4]<br>UWTUEMKLYAGTNQ-UPHRSURJSA-N | $2.4\times10^{-3}$ | | Yaffe et al. (2003) | Q | 248, 249 |
| 3-bromo-1-propene<br>$C_3H_5Br$<br>(allyl bromide)<br>[106-95-6]<br>BHELZAPQIKSEDF-UHFFFAOYSA-N | $1.7\times10^{-3}$<br>$9.0\times10^{-4}$<br>$3.9\times10^{-3}$<br>$1.6\times10^{-3}$<br>$8.6\times10^{-3}$<br>$2.6\times10^{-3}$<br>$1.0\times10^{-3}$<br>$1.7\times10^{-3}$<br>$1.7\times10^{-3}$<br>$1.7\times10^{-3}$ | | Yaws (2003)<br>HSDB (2015)<br>Gharagheizi et al. (2012)<br>Gharagheizi et al. (2010)<br>Hilal et al. (2008)<br>Modarresi et al. (2007)<br>Yao et al. (2002)<br>Yaws (1999)<br>Yaws and Yang (1992)<br>Abraham et al. (1990) | X<br>Q<br>Q<br>Q<br>Q<br>Q<br>Q<br>?<br>?<br>? | 237<br>99<br><br>246<br><br>67<br>229<br>21<br>21<br> |
| 3-bromo-1-propyne<br>$C_3H_3Br$<br>(propargyl bromide)<br>[106-96-7]<br>YORCIIVHUBAYBQ-UHFFFAOYSA-N | $8.8\times10^{-3}$<br>$1.6\times10^{-2}$<br>$8.4\times10^{-3}$<br><br>$8.7\times10^{-3}$<br> | 4000<br><br><br>3200<br><br>4200 | Yates and Gan (1998)<br>Keshavarz et al. (2022)<br>Duchowicz et al. (2020)<br>Kühne et al. (2005)<br>Duchowicz et al. (2020)<br>Kühne et al. (2005)<br>Fogg and Sangster (2003) | M<br>Q<br>Q<br>Q<br>?<br>?<br>W | 1<br><br><br><br>185, 21<br><br>788 |
| 1-bromocyclohexene<br>$C_6H_9Br$<br>[2044-08-8]<br>QBUMXSSCYUMVAW-UHFFFAOYSA-N | $2.0\times10^{-3}$ | | Hilal et al. (2008) | Q | |



Table A7.1: Bromocarbons (C, H, O, N, Br) (. . . continued)

| Substance Formula (Trivial Name) [CAS Registry Number] InChIKey | $H_s^{cp}$ (at $T^\ominus$) $\left[\dfrac{\mathrm{mol}}{\mathrm{m}^3\,\mathrm{Pa}}\right]$ | $\dfrac{\mathrm{d}\ln H_s^{cp}}{\mathrm{d}(1/T)}$ [K] | Reference | Type | Note |
|---|---|---|---|---|---|
| 1-bromo-4-methylcyclohexene $C_7H_{11}Br$ [31053-84-6] LIUFYLLBQSSKQS-UHFFFAOYSA-N | $1.4\times10^{-3}$ | | Hilal et al. (2008) | Q | |
| bromobenzene $C_6H_5Br$ [108-86-1] QARVLSVVCXYDNA-UHFFFAOYSA-N | $4.3\times10^{-3}$ | 4500 | Brockbank (2013) | L | 1 |
| | $5.0\times10^{-3}$ | 4200 | Fogg and Sangster (2003) | L | |
| | $4.8\times10^{-3}$ | | Mackay and Shiu (1981) | L | |
| | $6.0\times10^{-3}$ | 4300 | Hiatt (2013) | M | |
| | $3.9\times10^{-3}$ | 2900 | Lau et al. (2010) | M | 11 |
| | $5.0\times10^{-3}$ | | de Wolf and Lieder (1998) | M | 87 |
| | $4.0\times10^{-3}$ | | Shiu and Mackay (1997) | M | |
| | $6.1\times10^{-3}$ | | Hovorka and Dohnal (1997) | M | 12 |
| | $4.9\times10^{-3}$ | 4200 | Kondoh and Nakajima (1997) | M | |
| | $5.3\times10^{-3}$ | 5300 | Hansen et al. (1993) | M | 281 |
| | $4.4\times10^{-3}$ | | Li and Carr (1993) | M | |
| | $4.0\times10^{-3}$ | | Mackay and Shiu (1981) | M | |
| | $4.7\times10^{-3}$ | | Shiu and Mackay (1997) | V | |
| | $4.7\times10^{-3}$ | | Mackay et al. (1993) | V | |
| | $5.0\times10^{-3}$ | | Hwang et al. (1992) | V | |
| | $4.7\times10^{-3}$ | | Hine and Mookerjee (1975) | V | |
| | $4.6\times10^{-3}$ | | Yaws (2003) | X | 237 |
| | $4.7\times10^{-3}$ | | HSDB (2015) | C | |
| | $4.0\times10^{-3}$ | | Schüürmann (2000) | C | 21 |
| | $7.4\times10^{-3}$ | | Keshavarz et al. (2022) | Q | |
| | $7.9\times10^{-3}$ | | Duchowicz et al. (2020) | Q | 184 |
| | $3.1\times10^{-2}$ | | Gharagheizi et al. (2012) | Q | |
| | $6.2\times10^{-3}$ | | Raventos-Duran et al. (2010) | Q | 242, 243 |
| | $6.2\times10^{-3}$ | | Raventos-Duran et al. (2010) | Q | 244 |
| | $4.9\times10^{-3}$ | | Raventos-Duran et al. (2010) | Q | 245 |
| | $4.2\times10^{-3}$ | | Gharagheizi et al. (2010) | Q | 246 |
| | $5.2\times10^{-3}$ | | Hilal et al. (2008) | Q | |
| | $7.7\times10^{-3}$ | | Modarresi et al. (2007) | Q | 67 |
| | | 4800 | Kühne et al. (2005) | Q | |
| | $4.1\times10^{-3}$ | | Yaffe et al. (2003) | Q | 248, 249 |
| | $4.2\times10^{-3}$ | | Yao et al. (2002) | Q | 229 |
| | $5.4\times10^{-3}$ | | English and Carroll (2001) | Q | 230, 231 |
| | $2.9\times10^{-3}$ | | Katritzky et al. (1998) | Q | |
| | $5.1\times10^{-3}$ | | Suzuki et al. (1992) | Q | 232 |
| | $7.3\times10^{-3}$ | | Nirmalakhandan and Speece (1988) | Q | |
| | $4.0\times10^{-3}$ | | Duchowicz et al. (2020) | ? | 185, 21 |
| | | 4300 | Kühne et al. (2005) | ? | |
| | $4.6\times10^{-3}$ | | Yaws (1999) | ? | 21 |
| | $4.7\times10^{-3}$ | | Yaws and Yang (1992) | ? | 21 |
| | $4.7\times10^{-3}$ | | Abraham et al. (1990) | ? | |



Table A7.1: Bromocarbons (C, H, O, N, Br) (...continued)

| Substance Formula (Trivial Name) [CAS Registry Number] InChIKey | $H_s^{cp}$ (at $T^{\ominus}$) $\left[\dfrac{\mathrm{mol}}{\mathrm{m^3\,Pa}}\right]$ | $\dfrac{\mathrm{d\ln} H_s^{cp}}{\mathrm{d}(1/T)}$ [K] | Reference | Type | Note |
|---|---|---|---|---|---|
| bromobenzene-d5 $C_6D_5Br$ [4165-57-5] QARVLSVVCXYDNA-RALIUCGRSA-N | $6.5\times10^{-3}$ | 4200 | Hiatt (2013) | M | |
| 1,2-dibromobenzene $C_6H_4Br_2$ [583-53-9] WQONPSCCEXUXTQ-UHFFFAOYSA-N | $9.5\times10^{-3}$ | | Schüürmann (2000) | V | |
| 1,3-dibromobenzene $C_6H_4Br_2$ [108-36-1] JSRLURSZEMLAFO-UHFFFAOYSA-N | $1.2\times10^{-2}$ | | Brockbank (2013) | L | |
| | $8.0\times10^{-3}$ | | Duchowicz et al. (2020) | V | 186 |
| | $5.0\times10^{-3}$ | | Mackay and Shiu (1981) | V | 555 |
| | $1.2\times10^{-2}$ | | Yaws (2003) | X | 237 |
| | $7.7\times10^{-3}$ | | Duchowicz et al. (2020) | Q | |
| | $1.2\times10^{-2}$ | | Raventos-Duran et al. (2010) | Q | 242, 243 |
| | $2.0\times10^{-2}$ | | Raventos-Duran et al. (2010) | Q | 244 |
| | $1.2\times10^{-2}$ | | Raventos-Duran et al. (2010) | Q | 245 |
| | $1.2\times10^{-2}$ | | Gharagheizi et al. (2010) | Q | 246 |
| | $9.0\times10^{-3}$ | | Hilal et al. (2008) | Q | |
| | $6.4\times10^{-3}$ | | Modarresi et al. (2007) | Q | 67 |
| | $7.0\times10^{-3}$ | | Yao et al. (2002) | Q | 229 |
| | $4.2\times10^{-2}$ | | Katritzky et al. (1998) | Q | |
| | $1.2\times10^{-2}$ | | Yaws (1999) | ? | 21 |
| 1,4-dibromobenzene $C_6H_4Br_2$ [106-37-6] SWJPEBQEEAHIGZ-UHFFFAOYSA-N | $9.4\times10^{-3}$ | | Kuramochi et al. (2004) | M | |
| | $1.1\times10^{-2}$ | | Duchowicz et al. (2020) | V | 186 |
| | $1.1\times10^{-2}$ | | HSDB (2015) | V | |
| | $4.3\times10^{-3}$ | | Schüürmann (2000) | V | |
| | $4.8\times10^{-3}$ | | Mackay and Shiu (1981) | V | 555 |
| | $2.0\times10^{-2}$ | | Hine and Mookerjee (1975) | V | |
| | $1.1\times10^{-2}$ | | Kuramochi et al. (2004) | C | |
| | $7.1\times10^{-3}$ | | Duchowicz et al. (2020) | Q | |
| | $1.2\times10^{-2}$ | | Raventos-Duran et al. (2010) | Q | 242, 243 |
| | $2.5\times10^{-2}$ | | Raventos-Duran et al. (2010) | Q | 244 |
| | $1.2\times10^{-2}$ | | Raventos-Duran et al. (2010) | Q | 245 |
| | $1.2\times10^{-2}$ | | Hilal et al. (2008) | Q | |
| | $1.0\times10^{-2}$ | | Modarresi et al. (2007) | Q | 67 |
| | | 5600 | Kühne et al. (2005) | Q | |
| | $4.5\times10^{-3}$ | | English and Carroll (2001) | Q | 230, 274 |
| | $4.1\times10^{-2}$ | | Katritzky et al. (1998) | Q | |
| | $2.4\times10^{-2}$ | | Nirmalakhandan and Speece (1988) | Q | |
| | | 6900 | Kühne et al. (2005) | ? | |
| 1,2,4-tribromobenzene $C_6H_3Br_3$ [615-54-3] FWAJPSIPOULHHH-UHFFFAOYSA-N | $3.1\times10^{-2}$ | | Kuramochi et al. (2004) | M | |
| | $2.9\times10^{-2}$ | | Kuramochi et al. (2004) | C | |
| | $1.8\times10^{-2}$ | | Hilal et al. (2008) | Q | |



Table A7.1: Bromocarbons (C, H, O, N, Br) (...continued)

| Substance<br>Formula<br>(Trivial Name)<br>[CAS Registry Number]<br>InChIKey | $H_s^{cp}$<br>(at $T^{\ominus}$)<br>$\left[\dfrac{\text{mol}}{\text{m}^3\,\text{Pa}}\right]$ | $\dfrac{\text{d}\ln H_s^{cp}}{\text{d}(1/T)}$<br><br>[K] | Reference | Type | Note |
|---|---|---|---|---|---|
| 1,3,5-tribromobenzene<br>$C_6H_3Br_3$<br>[626-39-1]<br>YWDUZLFWHVQCHY-UHFFFAOYSA-N | $1.3\times10^{-2}$<br>$5.8\times10^{-3}$<br>$2.9\times10^{-2}$<br>$4.0\times10^{-2}$<br>$2.5\times10^{-2}$<br>$2.6\times10^{-2}$<br>$5.1\times10^{-1}$ | | Duchowicz et al. (2020)<br>Duchowicz et al. (2020)<br>Zhang et al. (2010)<br>Zhang et al. (2010)<br>Zhang et al. (2010)<br>Zhang et al. (2010)<br>Katritzky et al. (1998) | V<br>Q<br>Q<br>Q<br>Q<br>Q<br>Q | 186<br><br>287, 288<br>287, 289<br>287, 290<br>287, 291<br> |
| 1,2,4,5-tetrabromobenzene<br>$C_6H_2Br_4$<br>[636-28-2]<br>QCKHVNQHBOGZER-UHFFFAOYSA-N | $2.7\times10^{-3}$<br>$2.0\times10^{-2}$ | | Kuramochi et al. (2004)<br>Hilal et al. (2008) | M<br>Q | |
| hexabromobenzene<br>$C_6Br_6$<br>[87-82-1]<br>CAYGQBVSOZLICD-UHFFFAOYSA-N | $9.3\times10^{-2}$<br>$4.1\times10^{-1}$<br>$7.1$<br>$3.5\times10^{-1}$<br>$4.0\times10^{-1}$<br>$4.6\times10^{-1}$<br>$4.6\times10^{-1}$<br>$6.0\times10^{-2}$<br>$6.7\times10^{-1}$<br>$1.2\times10^{-2}$ | | Kuramochi et al. (2004)<br>Kuramochi et al. (2014)<br>Tittlemier et al. (2002)<br>HSDB (2015)<br>Xiao et al. (2012)<br>Zhang et al. (2010)<br>Zhang et al. (2010)<br>Zhang et al. (2010)<br>Zhang et al. (2010)<br>Hilal et al. (2008) | M<br>V<br>V<br>Q<br>Q<br>Q<br>Q<br>Q<br>Q<br>Q | <br><br><br>99<br><br>287, 288<br>287, 289<br>287, 290<br>287, 291<br> |
| (bromomethyl)-benzene<br>$C_7H_7Br$<br>(benzyl bromide)<br>[100-39-0]<br>AGEZXYOZHKGVCM-UHFFFAOYSA-N | $1.4\times10^{-3}$<br>$5.4\times10^{-2}$<br>$2.2\times10^{-2}$ | | HSDB (2015)<br>Hilal et al. (2008)<br>Abraham et al. (1990) | Q<br>Q<br>? | 99<br><br> |
| $p$-bromobenzyl bromide<br>$C_7H_6Br_2$<br>[589-15-1]<br>YLRBJYMANQKEAW-UHFFFAOYSA-N | $3.6\times10^{-2}$<br>$2.7\times10^{-1}$<br>$2.0\times10^{-1}$<br>$2.4\times10^{-2}$ | | Zhang et al. (2010)<br>Zhang et al. (2010)<br>Zhang et al. (2010)<br>Zhang et al. (2010) | Q<br>Q<br>Q<br>Q | 287, 288<br>287, 289<br>287, 290<br>287, 291 |
| 1-bromo-2-methylbenzene<br>$BrC_6H_4CH_3$<br>($o$-bromotoluene)<br>[95-46-5]<br>QSSXJPIWXQTSIX-UHFFFAOYSA-N | $4.1\times10^{-3}$<br>$5.3\times10^{-3}$ | | HSDB (2015)<br>Hilal et al. (2008) | Q<br>Q | 99<br> |
| 1-bromo-3-methylbenzene<br>$BrC_6H_4CH_3$<br>($m$-bromotoluene)<br>[591-17-3]<br>WJIFKOVZNJTSGO-UHFFFAOYSA-N | $1.5\times10^{-3}$<br>$1.5\times10^{-3}$<br>$4.0\times10^{-3}$<br>$5.2\times10^{-3}$<br>$4.8\times10^{-3}$<br>$1.5\times10^{-3}$<br>$7.9\times10^{-3}$ | | Duchowicz et al. (2020)<br>HSDB (2015)<br>Duchowicz et al. (2020)<br>Hilal et al. (2008)<br>Modarresi et al. (2007)<br>Yaffe et al. (2003)<br>Katritzky et al. (1998) | V<br>V<br>Q<br>Q<br>Q<br>Q<br>Q | 186<br><br><br><br>67<br>248, 249<br> |



Table A7.1: Bromocarbons (C, H, O, N, Br) (...continued)

| Substance Formula (Trivial Name) [CAS Registry Number] InChIKey | $H_s^{cp}$ (at $T^{\ominus}$) $\left[\dfrac{\text{mol}}{\text{m}^3\,\text{Pa}}\right]$ | $\dfrac{\text{d}\ln H_s^{cp}}{\text{d}(1/T)}$ [K] | Reference | Type | Note |
|---|---|---|---|---|---|
| 1-bromo-4-methylbenzene | $3.6\times10^{-3}$ | 4800 | Brockbank (2013) | L | 1 |
| BrC$_6$H$_4$CH$_3$ | $3.4\times10^{-3}$ | 4600 | Brockbank et al. (2013) | M | |
| (p-bromotoluene) | $4.2\times10^{-3}$ | | Hine and Mookerjee (1975) | V | |
| [106-38-7] | $5.3\times10^{-2}$ | | Keshavarz et al. (2022) | Q | |
| ZBTMRBYMKUEVEU-UHFFFAOYSA-N | $4.0\times10^{-3}$ | | Duchowicz et al. (2020) | Q | 299 |
| | $4.2\times10^{-3}$ | | Li et al. (2014) | Q | 241 |
| | $5.6\times10^{-3}$ | | Hilal et al. (2008) | Q | |
| | $5.5\times10^{-3}$ | | Modarresi et al. (2007) | Q | 67 |
| | $4.5\times10^{-3}$ | | Yaffe et al. (2003) | Q | 248, 249 |
| | $3.4\times10^{-3}$ | | English and Carroll (2001) | Q | 230, 231 |
| | $8.0\times10^{-3}$ | | Katritzky et al. (1998) | Q | |
| | $5.2\times10^{-3}$ | | Nirmalakhandan et al. (1997) | Q | |
| | $3.6\times10^{-3}$ | | Suzuki et al. (1992) | Q | 232 |
| | $5.2\times10^{-3}$ | | Nirmalakhandan and Speece (1988) | Q | |
| | $4.2\times10^{-3}$ | | Duchowicz et al. (2020) | ? | 185, 21 |
| | $4.2\times10^{-3}$ | | Abraham et al. (1990) | ? | |
| 3,5-dibromotoluene | $1.7\times10^{-2}$ | 4800 | Hiatt (2013) | M | |
| C$_7$H$_6$Br$_2$ | | | | | |
| [1611-92-3] | | | | | |
| DPKKOVGCHDUSAI-UHFFFAOYSA-N | | | | | |
| pentabromotoluene | $4.0\times10^{-1}$ | | Xiao et al. (2012) | Q | |
| C$_7$H$_3$Br$_5$ | | | | | |
| [87-83-2] | | | | | |
| OZHJEQVYCBTHJT-UHFFFAOYSA-N | | | | | |
| 1-bromo-2-ethylbenzene | $3.0\times10^{-3}$ | | Hine and Mookerjee (1975) | V | |
| C$_8$H$_9$Br | $4.3\times10^{-3}$ | | Hilal et al. (2008) | Q | |
| [1973-22-4] | $2.8\times10^{-3}$ | | English and Carroll (2001) | Q | 230, 231 |
| HVRUGFJYCAFAAN-UHFFFAOYSA-N | $2.9\times10^{-3}$ | | Suzuki et al. (1992) | Q | 232 |
| | $4.5\times10^{-3}$ | | Nirmalakhandan and Speece (1988) | Q | |
| 1-bromo-4-ethylbenzene | $3.1\times10^{-3}$ | | Zhang et al. (2010) | Q | 287, 288 |
| C$_8$H$_9$Br | $6.1\times10^{-3}$ | | Zhang et al. (2010) | Q | 287, 289 |
| [1585-07-5] | $4.2\times10^{-3}$ | | Zhang et al. (2010) | Q | 287, 290 |
| URFPRAHGGBYNPW-UHFFFAOYSA-N | $5.2\times10^{-3}$ | | Zhang et al. (2010) | Q | 287, 291 |
| 1-(bromomethyl)-2-methylbenzene | $1.3\times10^{-2}$ | | HSDB (2015) | Q | 99 |
| C$_8$H$_9$Br | | | | | |
| (o-xylyl bromide) | | | | | |
| [89-92-9] | | | | | |
| WGVYCXYGPNNUQA-UHFFFAOYSA-N | | | | | |
| 1-(bromomethyl)-3-methylbenzene | $1.3\times10^{-2}$ | | HSDB (2015) | Q | 99 |
| C$_8$H$_9$Br | | | | | |
| (m-xylyl bromide) | | | | | |
| [620-13-3] | | | | | |
| FWLWTILKTABGKQ-UHFFFAOYSA-N | | | | | |



Table A7.1: Bromocarbons (C, H, O, N, Br) (...continued)

| Substance Formula (Trivial Name) [CAS Registry Number] InChIKey | $H_s^{cp}$ (at $T^\ominus$) $\left[\dfrac{\mathrm{mol}}{\mathrm{m^3\,Pa}}\right]$ | $\dfrac{\mathrm{d}\ln H_s^{cp}}{\mathrm{d}(1/T)}$ [K] | Reference | Type | Note |
|---|---|---|---|---|---|
| 1-(bromomethyl)-4-methylbenzene $C_8H_9Br$ (*p*-xylyl bromide) [104-81-4] WZRKSPFYXUXINF-UHFFFAOYSA-N | $1.3\times10^{-2}$ | | HSDB (2015) | Q | 99 |
| (2-bromoethyl)-benzene $C_8H_9Br$ [103-63-9] WMPPDTMATNBGJN-UHFFFAOYSA-N | $6.5\times10^{-3}$ $6.5\times10^{-3}$ $7.9\times10^{-3}$ $2.2\times10^{-2}$ $6.9\times10^{-3}$ $1.3\times10^{-2}$ | | Duchowicz et al. (2020) HSDB (2015) Duchowicz et al. (2020) Modarresi et al. (2007) Yaffe et al. (2003) Katritzky et al. (1998) | V V Q Q Q Q | 186 67 248, 249 |
| 2-bromostyrene $C_8H_7Br$ [125904-11-2] SSZOCHFYWWVSAI-UHFFFAOYSA-N | $9.0\times10^{-3}$ $9.5\times10^{-3}$ $7.3\times10^{-3}$ $1.5\times10^{-2}$ | | Zhang et al. (2010) Zhang et al. (2010) Zhang et al. (2010) Zhang et al. (2010) | Q Q Q Q | 287, 288 287, 289 287, 290 287, 291 |
| (2-bromoethenyl)benzene $C_8H_7Br$ [103-64-0] YMOONIIMQBGTDU-VOTSOKGWSA-N | $1.8\times10^{-2}$ | | HSDB (2015) | Q | 99 |
| 2,3,4,5,6-pentabromoethylbenzene $C_8H_5Br_5$ [85-22-3] FIAXCDIQXHJNIX-UHFFFAOYSA-N | $1.2\times10^{-1}$ $3.6\times10^{-1}$ $3.3\times10^{-2}$ $9.7\times10^{-2}$ | | Zhang et al. (2010) Zhang et al. (2010) Zhang et al. (2010) Zhang et al. (2010) | Q Q Q Q | 287, 288 287, 289 287, 290 287, 291 |
| 1-bromo-2-(2-propyl)-benzene $BrC_6H_4C_3H_7$ (*o*-bromocumene) [7073-94-1] LECYCYNAEJDSIL-UHFFFAOYSA-N | $1.7\times10^{-3}$ $2.5\times10^{-3}$ $2.0\times10^{-3}$ $3.1\times10^{-3}$ | | Hine and Mookerjee (1975) Hilal et al. (2008) Suzuki et al. (1992) Nirmalakhandan and Speece (1988) | V Q Q Q | 232 |
| 1-bromonaphthalene $C_{10}H_7Br$ [90-11-9] DLKQHBOKULLWDQ-UHFFFAOYSA-N | $3.5\times10^{-2}$ $5.0\times10^{-2}$ $3.0\times10^{-2}$ $5.0\times10^{-2}$ $8.2\times10^{-2}$ $3.7\times10^{-2}$ $1.8\times10^{-2}$ $3.5\times10^{-2}$ | | Duchowicz et al. (2020) Yaws (2003) Duchowicz et al. (2020) Gharagheizi et al. (2010) Hilal et al. (2008) Yaffe et al. (2003) Katritzky et al. (1998) Yaws (1999) | V X Q Q Q Q Q ? | 186 237, 294 246 248, 249 21, 294 |
| 2-bromonaphthalene $C_{10}H_7Br$ [580-13-2] APSMUYYLXZULMS-UHFFFAOYSA-N | $4.3\times10^{-2}$ | | Ebert et al. (2023) | ? | 318 |



Table A7.1: Bromocarbons (C, H, O, N, Br) (. . . continued)

| Substance<br>Formula<br>(Trivial Name)<br>[CAS Registry Number]<br>InChIKey | $H_s^{cp}$<br>(at $T^\ominus$)<br><br>$\left[\dfrac{\text{mol}}{\text{m}^3\,\text{Pa}}\right]$ | $\dfrac{\text{d}\ln H_s^{cp}}{\text{d}(1/T)}$<br><br>[K] | Reference | Type | Note |
|---|---|---|---|---|---|
| 1,4-dibromonaphthalene<br>$C_{10}H_6Br_2$<br>[83-53-4]<br>IBGUDZMIAZLJNY-UHFFFAOYSA-N | $5.8\times10^{-2}$ | | Ebert et al. (2023) | ? | 316 |
| decabromobiphenyl<br>$C_{12}Br_{10}$<br>[13654-09-6]<br>AQPHBYQUCKHJLT-UHFFFAOYSA-N | $2.3\times10^2$<br>$2.4\times10^2$<br>$3.0\times10^2$<br>$2.3\times10^2$<br>$5.0\times10^2$ | | HSDB (2015)<br>Zhang et al. (2010)<br>Zhang et al. (2010)<br>Zhang et al. (2010)<br>Zhang et al. (2010) | Q<br>Q<br>Q<br>Q<br>Q | 99<br>287, 288<br>287, 289<br>287, 290<br>287, 291 |
| 4-bromo-1,1'-biphenyl<br>$C_{12}H_9Br$<br>[92-66-0]<br>PKJBWOWQJHHAHG-UHFFFAOYSA-N | $6.0\times10^{-2}$<br>$6.9\times10^{-2}$<br>$1.7\times10^{-1}$<br>$3.5\times10^{-1}$ | | Zhang et al. (2010)<br>Zhang et al. (2010)<br>Zhang et al. (2010)<br>Zhang et al. (2010) | Q<br>Q<br>Q<br>Q | 287, 288<br>287, 289<br>287, 290<br>287, 291 |
| octabromobiphenyl<br>$C_{12}H_2Br_8$<br>[27858-07-7]<br>NDRKXFLZSRHAJE-UHFFFAOYSA-N | $4.1\times10^3$ | | HSDB (2015) | V | |
| 2,2',4,4',5,5'-hexabromobiphenyl<br>$C_{12}H_4Br_6$<br>[59080-40-9]<br>HMBBJSKXDBUNNT-UHFFFAOYSA-N | $2.3$ | | HSDB (2015) | V | |
| 1,2-bis(pentabromophenyl) ethane<br>$C_{14}H_4Br_{10}$<br>[84852-53-9]<br>BZQKBFHEWDPQHD-UHFFFAOYSA-N | $1.5\times10^2$<br>$8.8\times10^2$<br>$8.6\times10^1$<br>$1.7\times10^2$ | | Zhang et al. (2010)<br>Zhang et al. (2010)<br>Zhang et al. (2010)<br>Zhang et al. (2010) | Q<br>Q<br>Q<br>Q | 287, 288<br>287, 289<br>287, 290<br>287, 291 |
| 1,3,6,8-tetrabromopyrene<br>$C_{16}H_6Br_4$<br>[128-63-2]<br>ZKBKRTZIYOKNRG-UHFFFAOYSA-N | $4.7\times10^1$<br>$4.4\times10^1$<br>$6.2\times10^{-1}$<br>$6.5\times10^1$ | | Zhang et al. (2010)<br>Zhang et al. (2010)<br>Zhang et al. (2010)<br>Zhang et al. (2010) | Q<br>Q<br>Q<br>Q | 287, 288<br>287, 289<br>287, 290<br>287, 291 |
| 6-bromobenzo[$a$]pyrene<br>$C_{20}H_{11}Br$<br>[21248-00-0]<br>MJSYSGSEEADMTK-UHFFFAOYSA-N | $1.2\times10^2$ | | HSDB (2015) | Q | 99 |
| bromomethanol<br>$CH_2BrOH$<br>[50398-29-3]<br>OEDMOCYNWLHUDP-UHFFFAOYSA-N | $4.1$<br>$3.2\times10^1$<br>$2.5\times10^2$<br>$2.0\times10^1$ | | Wang et al. (2017)<br>Wang et al. (2017)<br>Wang et al. (2017)<br>Krysztofiak et al. (2012) | Q<br>Q<br>Q<br>Q | 80, 238<br>80, 239<br>80, 240<br> |
| dibromomethanol<br>$CHBr_2OH$<br>[166600-78-8]<br>WICMSCLXMMZMFF-UHFFFAOYSA-N | $1.7\times10^2$ | | Krysztofiak et al. (2012) | Q | |





Table A7.1: Bromocarbons (C, H, O, N, Br) (... continued)

| Substance<br>Formula<br>(Trivial Name)<br>[CAS Registry Number]<br>InChIKey | $H_s^{cp}$<br>(at $T^{\ominus}$)<br>$\left[\dfrac{\mathrm{mol}}{\mathrm{m^3\,Pa}}\right]$ | $\dfrac{\mathrm{d}\ln H_s^{cp}}{\mathrm{d}(1/T)}$<br><br>[K] | Reference | Type | Note |
|---|---|---|---|---|---|
| tribromomethanol<br>CBr$_3$OH<br>[5405-30-1]<br>ACRXLLXANWELLX-UHFFFAOYSA-N | $1.5\times10^3$ | | Krysztofiak et al. (2012) | Q | |
| formyl bromide<br>CHBrO<br>[7726-11-6]<br>AIFARXRIYKCEEV-UHFFFAOYSA-N | $4.5\times10^{-1}$<br>$3.2\times10^{-2}$<br>$3.7\times10^{-3}$<br>$7.3\times10^{-1}$ | | Wang et al. (2017)<br>Wang et al. (2017)<br>Wang et al. (2017)<br>Krysztofiak et al. (2012) | Q<br>Q<br>Q<br>Q | 80, 238<br>80, 239<br>80, 240<br> |
| carbonyl bromide<br>CBr$_2$O<br>[593-95-3]<br>MOIPGXQKZSZOQX-UHFFFAOYSA-N | $2.1\times10^{-1}$ | | Krysztofiak et al. (2012) | Q | |
| bromomethyl peroxide<br>CH$_2$BrO$_2$H<br>WXVCUAGYVVWQBP-UHFFFAOYSA-N | $3.7\times10^1$<br>$1.3\times10^2$<br>$3.0\times10^1$<br>$2.5\times10^1$ | | Wang et al. (2017)<br>Wang et al. (2017)<br>Wang et al. (2017)<br>Krysztofiak et al. (2012) | Q<br>Q<br>Q<br>Q | 80, 238<br>80, 239<br>80, 240<br> |
| dibromomethyl peroxide<br>CHBr$_2$O$_2$H<br>JBKQCSQVMCXDPZ-UHFFFAOYSA-N | $2.2\times10^2$ | | Krysztofiak et al. (2012) | Q | |
| tribromomethyl peroxide<br>CBr$_3$O$_2$H<br>WJYKCADCZNTKBI-UHFFFAOYSA-N | $1.9\times10^3$ | | Krysztofiak et al. (2012) | Q | |
| bromoethanoic acid<br>CH$_2$BrCOOH<br>(bromoacetic acid)<br>[79-08-3]<br>KDPAWGWELVVRCH-UHFFFAOYSA-N | $1.5\times10^3$<br>$1.5\times10^3$<br>$1.5\times10^3$<br>$1.5\times10^3$<br>$5.6\times10^2$<br>$1.5\times10^2$<br>$4.2\times10^2$<br>$5.6\times10^3$<br>$5.5\times10^2$<br>$2.0\times10^3$<br>$2.5\times10^3$<br>$1.6\times10^2$<br><br>$1.5\times10^3$<br> | 9300<br>9300<br>9300<br>9300<br><br><br><br><br><br><br><br><br>8800<br><br>9300 | Burkholder et al. (2019)<br>Burkholder et al. (2015)<br>Sander et al. (2011)<br>Bowden et al. (1998a)<br>Keshavarz et al. (2022)<br>Duchowicz et al. (2020)<br>Wang et al. (2017)<br>Wang et al. (2017)<br>Wang et al. (2017)<br>Raventos-Duran et al. (2010)<br>Raventos-Duran et al. (2010)<br>Raventos-Duran et al. (2010)<br>Kühne et al. (2005)<br>Duchowicz et al. (2020)<br>Kühne et al. (2005) | L<br>L<br>L<br>M<br>Q<br>Q<br>Q<br>Q<br>Q<br>Q<br>Q<br>Q<br>Q<br>?<br>? | <br><br><br><br><br>184<br>80, 238<br>80, 239<br>80, 240<br>242, 243<br>244<br>245<br><br>185, 21<br> |
| dibromoethanoic acid<br>CHBr$_2$COOH<br>(dibromoacetic acid)<br>[631-64-1]<br>SIEILFNCEFEENQ-UHFFFAOYSA-N | $2.3\times10^3$<br>$2.3\times10^3$<br>$2.3\times10^3$<br>$2.2\times10^3$<br>$5.6\times10^2$<br>$4.9\times10^2$<br>$3.9\times10^3$ | 8900<br>8900<br>8900<br>8900 | Burkholder et al. (2019)<br>Burkholder et al. (2015)<br>Sander et al. (2011)<br>Bowden et al. (1998a)<br>Keshavarz et al. (2022)<br>Duchowicz et al. (2020)<br>Raventos-Duran et al. (2010) | L<br>L<br>L<br>M<br>Q<br>Q<br>Q | <br><br><br><br><br>184<br>242, 243 |



Table A7.1: Bromocarbons (C, H, O, N, Br) (... continued)

| Substance Formula (Trivial Name) [CAS Registry Number] InChIKey | $H_s^{cp}$ (at $T^{\ominus}$) $\left[\dfrac{\text{mol}}{\text{m}^3\,\text{Pa}}\right]$ | $\dfrac{\text{d}\ln H_s^{cp}}{\text{d}(1/T)}$ [K] | Reference | Type | Note |
|---|---|---|---|---|---|
| | $6.2\times10^2$ | | Raventos-Duran et al. (2010) | Q | 244 |
| | $1.2\times10^3$ | | Raventos-Duran et al. (2010) | Q | 245 |
| | | 9900 | Kühne et al. (2005) | Q | |
| | $2.2\times10^3$ | | Duchowicz et al. (2020) | ? | 185, 21 |
| | | 9000 | Kühne et al. (2005) | ? | |
| tribromoethanoic acid $CBr_3COOH$ (tribromoacetic acid) [75-96-7] QIONYIKHPASLHO-UHFFFAOYSA-N | $3.0\times10^3$ | 9000 | Burkholder et al. (2019) | L | |
| | $3.0\times10^3$ | 9000 | Burkholder et al. (2015) | L | |
| | $3.0\times10^3$ | 9000 | Sander et al. (2011) | L | |
| | $2.9\times10^3$ | 9000 | Bowden et al. (1998a) | M | |
| | $5.6\times10^2$ | | Keshavarz et al. (2022) | Q | |
| | $3.5\times10^2$ | | Duchowicz et al. (2020) | Q | |
| | $2.0\times10^3$ | | Raventos-Duran et al. (2010) | Q | 271, 243 |
| | 9.9 | | Raventos-Duran et al. (2010) | Q | 244 |
| | $1.2\times10^4$ | | Raventos-Duran et al. (2010) | Q | 245 |
| | $3.0\times10^3$ | | Duchowicz et al. (2020) | ? | 185, 21 |
| MCM:BRETO3H $C_2H_3O_3Br$ RGXORGOPJDPEIK-UHFFFAOYSA-N | $4.5\times10^2$ | | Wang et al. (2017) | Q | 80, 238 |
| | $1.4\times10^2$ | | Wang et al. (2017) | Q | 80, 239 |
| | $9.8\times10^{-1}$ | | Wang et al. (2017) | Q | 80, 240 |
| MCM:DIBRETO2H $C_2H_4O_2Br_2$ ZXSZEFURCAPOPD-UHFFFAOYSA-N | $5.4\times10^2$ | | Wang et al. (2017) | Q | 80, 238 |
| | $2.2\times10^2$ | | Wang et al. (2017) | Q | 80, 239 |
| | $9.6\times10^1$ | | Wang et al. (2017) | Q | 80, 240 |
| MCM:DIBRETOH $C_2H_4OBr_2$ IYJYRKYSUOTLAE-UHFFFAOYSA-N | $1.1\times10^2$ | | Wang et al. (2017) | Q | 80, 238 |
| | $1.1\times10^2$ | | Wang et al. (2017) | Q | 80, 239 |
| | $1.7\times10^3$ | | Wang et al. (2017) | Q | 80, 240 |
| MCM:BRETAL $C_2H_3OBr$ NMPVEAUIHMEAQP-UHFFFAOYSA-N | $4.1\times10^{-1}$ | | Wang et al. (2017) | Q | 80, 238 |
| | 6.0 | | Wang et al. (2017) | Q | 80, 239 |
| | $2.5\times10^{-1}$ | | Wang et al. (2017) | Q | 80, 240 |
| 3-bromopropanol $C_3H_7BrO$ [627-18-9] RQFUZUMFPRMVDX-UHFFFAOYSA-N | 8.2 | | Modarresi et al. (2007) | Q | 67 |
| 2,3-dibromopropanol $C_3H_6Br_2O$ [96-13-9] QWVCIORZLNBIIC-UHFFFAOYSA-N | $1.6\times10^2$ | | HSDB (2015) | Q | 99 |
| | $1.6\times10^2$ | | Zhang et al. (2010) | Q | 287, 288 |
| | $1.1\times10^2$ | | Zhang et al. (2010) | Q | 287, 289 |
| | $1.2\times10^1$ | | Zhang et al. (2010) | Q | 287, 290 |
| | $1.4\times10^1$ | | Zhang et al. (2010) | Q | 287, 291 |
| bromoacetone $C_3H_5BrO$ [598-31-2] VQFAIAKCILWQPZ-UHFFFAOYSA-N | 1.7 | | HSDB (2015) | Q | 99 |



Table A7.1: Bromocarbons (C, H, O, N, Br) (... continued)

| Substance Formula (Trivial Name) [CAS Registry Number] InChIKey | $H_s^{cp}$ (at $T^\ominus$) $\left[\dfrac{\mathrm{mol}}{\mathrm{m}^3\,\mathrm{Pa}}\right]$ | $\dfrac{\mathrm{d}\ln H_s^{cp}}{\mathrm{d}(1/T)}$ [K] | Reference | Type | Note |
|---|---|---|---|---|---|
| (bromomethyl)oxirane C$_3$H$_5$BrO (epibromohydrin) [3132-64-7] GKIPXFAANLTWBM-UHFFFAOYSA-N | 4.1 | | HSDB (2015) | Q | 99 |
| 2,3-dibromobutane-1,4-diol C$_4$H$_8$Br$_2$O$_2$ [90801-18-6] OXYNQEOLHRWEPE-QWWZWVQMSA-N | $3.2\times10^3$ $1.0\times10^5$ $1.5\times10^6$ $4.7\times10^4$ | | Zhang et al. (2010) Zhang et al. (2010) Zhang et al. (2010) Zhang et al. (2010) | Q Q Q Q | 287, 288 287, 289 287, 290 287, 291 |
| bromoacetic acid, ethyl ester C$_4$H$_7$BrO$_2$ [105-36-2] PQJJJMRNHATNKG-UHFFFAOYSA-N | $3.7\times10^{-1}$ | | HSDB (2015) | Q | 99 |
| brometone C$_4$H$_7$Br$_3$O (1,1,1-tribromo-2-methyl-2-propanol) [76-08-4] JUGRTVJQTFZHOM-UHFFFAOYSA-N | $1.0\times10^3$ | | HSDB (2015) | Q | 99 |
| 2,2-bis(bromomethyl)-1,3-propanediol C$_5$H$_{10}$Br$_2$O$_2$ [3296-90-0] CHUGKEQJSLOLHL-UHFFFAOYSA-N | $2.4\times10^3$ | | HSDB (2015) | Q | 99 |
| trisbromoneopentyl alcohol C$_5$H$_9$Br$_3$O [36483-57-5] HQWKMYFWGMCJSW-UHFFFAOYSA-N | $7.7\times10^2$ 7.5 $1.1\times10^1$ 1.0 | | Zhang et al. (2010) Zhang et al. (2010) Zhang et al. (2010) Zhang et al. (2010) | Q Q Q Q | 287, 288 287, 289 287, 290 287, 291 |
| 2-bromophenol HOC$_6$H$_4$Br [95-56-7] VADKRMSMGWJZCF-UHFFFAOYSA-N | $4.5\times10^1$ 4.2 | | HSDB (2015) Hilal et al. (2008) | Q Q | 99 |
| 3-bromophenol HOC$_6$H$_4$Br [591-20-8] MNOJRWOWILAHAV-UHFFFAOYSA-N | $4.5\times10^1$ $2.3\times10^1$ | | HSDB (2015) Hilal et al. (2008) | Q Q | 99 |
| 4-bromophenol HOC$_6$H$_4$Br [106-41-2] GZFGOTFRPZRKDS-UHFFFAOYSA-N | $6.7\times10^1$ $6.8\times10^1$ 4.3 $1.5\times10^2$ $6.5\times10^1$ $2.0\times10^1$ $2.5\times10^1$ | 8200 | Abraham et al. (1994a) Parsons et al. (1971) Keshavarz et al. (2022) Duchowicz et al. (2020) Li et al. (2014) Raventos-Duran et al. (2010) Raventos-Duran et al. (2010) | R T Q Q Q Q Q | 417 241 242, 243 244 |





Table A7.1: Bromocarbons (C, H, O, N, Br) (...continued)

| Substance Formula (Trivial Name) [CAS Registry Number] InChIKey | $H_s^{cp}$ (at $T^\ominus$) $\left[\dfrac{\mathrm{mol}}{\mathrm{m}^3\,\mathrm{Pa}}\right]$ | $\dfrac{\mathrm{d}\ln H_s^{cp}}{\mathrm{d}(1/T)}$ [K] | Reference | Type | Note |
|---|---|---|---|---|---|
| | $4.9\times10^1$ | | Raventos-Duran et al. (2010) | Q | 245 |
| | $1.6\times10^1$ | | Hilal et al. (2008) | Q | |
| | $2.3\times10^1$ | | Modarresi et al. (2007) | Q | 67 |
| | $6.5\times10^1$ | | Yaffe et al. (2003) | Q | 248, 249 |
| | $3.5\times10^1$ | | English and Carroll (2001) | Q | 230, 231 |
| | $1.3\times10^2$ | | Katritzky et al. (1998) | Q | |
| | $3.0\times10^2$ | | Nirmalakhandan et al. (1997) | Q | |
| | $3.3\times10^1$ | | Nirmalakhandan and Speece (1988) | Q | |
| | $6.5\times10^1$ | | Duchowicz et al. (2020) | ? | 185, 21 |
| | $6.9\times10^1$ | | Abraham et al. (1990) | ? | |
| 2,4-dibromophenol C$_6$H$_4$Br$_2$O [615-58-7] FAXWFCTVSHEODL-UHFFFAOYSA-N | $1.1\times10^2$ | | HSDB (2015) | Q | 99 |
| 2,6-dibromophenol C$_6$H$_4$Br$_2$O [608-33-3] SSIZLKDLDKIHEV-UHFFFAOYSA-N | $1.1\times10^2$ | | HSDB (2015) | Q | 99 |
| 2,4,6-tribromophenol C$_6$H$_3$Br$_3$O [118-79-6] BSWWXRFVMJHFBN-UHFFFAOYSA-N | $2.1\times10^2$ $2.8\times10^2$ $1.5\times10^{-1}$ $6.2$ $7.7$ | | HSDB (2015) Zhang et al. (2010) Zhang et al. (2010) Zhang et al. (2010) Zhang et al. (2010) | Q Q Q Q Q | 99 287, 288 287, 289 287, 290 287, 291 |
| 2,3,4,6-tetrabromophenol C$_6$H$_2$Br$_4$O [14400-94-3] CXPJZISGVIVNEL-UHFFFAOYSA-N | $7.0\times10^2$ | | HSDB (2015) | Q | 99 |
| pentabromophenol C$_6$HBr$_5$O [608-71-9] SVHOVVJFOWGYJO-UHFFFAOYSA-N | $1.8\times10^3$ $1.8\times10^3$ $1.2$ $2.2\times10^1$ $1.3\times10^2$ | | HSDB (2015) Zhang et al. (2010) Zhang et al. (2010) Zhang et al. (2010) Zhang et al. (2010) | Q Q Q Q Q | 99 287, 288 287, 289 287, 290 287, 291 |
| 1-bromo-2-methoxybenzene C$_7$H$_7$BrO (2-bromoanisole) [578-57-4] HTDQSWDEWGSAMN-UHFFFAOYSA-N | $2.9\times10^{-2}$ | | Pfeifer et al. (2001) | M | 731 |
| 1-bromo-3-methoxybenzene C$_7$H$_7$BrO (3-bromoanisole) [2398-37-0] PLDWAJLZAAHOGG-UHFFFAOYSA-N | $7.2\times10^{-3}$ | | Pfeifer et al. (2001) | M | 731 |



Table A7.1: Bromocarbons (C, H, O, N, Br) (...continued)

| Substance Formula (Trivial Name) [CAS Registry Number] InChIKey | $H_s^{cp}$ (at $T^{\ominus}$) $\left[\dfrac{\mathrm{mol}}{\mathrm{m^3\,Pa}}\right]$ | $\dfrac{\mathrm{d}\ln H_s^{cp}}{\mathrm{d}(1/T)}$ [K] | Reference | Type | Note |
|---|---|---|---|---|---|
| 1-bromo-4-methoxybenzene $C_7H_7BrO$ (4-bromoanisole) [104-92-7] QJPJQTDYNZXKQF-UHFFFAOYSA-N | $1.1\times10^{-2}$ | | Pfeifer et al. (2001) | M | 731 |
| 1,5-dibromo-2-methoxybenzene $C_7H_6Br_2O$ (2,4-dibromoanisole) [21702-84-1] XGXUGXPKRBQINS-UHFFFAOYSA-N | $8.1\times10^{-2}$ | | Pfeifer et al. (2001) | M | 731 |
| 1,3-dibromo-2-methoxybenzene $C_7H_6Br_2O$ (2,6-dibromoanisole) [38603-09-7] BMZVDHQOAJUZJL-UHFFFAOYSA-N | $3.7\times10^{-2}$ | | Pfeifer et al. (2001) | M | 731 |
| 1,2,3-tribromo-4-methoxybenzene $C_7H_5Br_3O$ (2,3,4-tribromoanisole) [95970-13-1] NKWYZAWFQZLPSU-UHFFFAOYSA-N | $9.9\times10^{-3}$ | | Ebert et al. (2023) | ? | 789 |
| 1,3,4-tribromo-2-methoxybenzene $C_7H_5Br_3O$ (2,3,6-tribromoanisole) [95970-19-7] XZYCSFIRWAPGJV-UHFFFAOYSA-N | $5.2\times10^{-3}$ | 2800 | Diaz et al. (2005) | M | 790 |
| 1,3,5-tribromo-2-methoxybenzene $C_7H_5Br_3O$ (2,4,6-tribromoanisole) [607-99-8] YXTRCOAFNXQTKL-UHFFFAOYSA-N | $1.9\times10^{-2}$ $1.3\times10^{-2}$ $3.1\times10^{-2}$ | 6400 | Diaz et al. (2005) Pfeifer et al. (2001) HSDB (2015) | M M Q | 731 99 |
| pentabromomethoxybenzene $C_7H_3Br_5O$ (pentabromoanisole) [1825-26-9] VEFNQGLEDJPUJF-UHFFFAOYSA-N | $1.0$ | | Pfeifer et al. (2001) | M | 731 |
| 1,3,5-tribromo-2-methoxy-4-methylbenzene $C_8H_7Br_3O$ [41424-36-6] NMPPAMFEYWGNCI-UHFFFAOYSA-N | $4.4\times10^{-1}$ $2.0\times10^{-1}$ $3.2\times10^{-1}$ $1.9\times10^{-1}$ | | Zhang et al. (2010) Zhang et al. (2010) Zhang et al. (2010) Zhang et al. (2010) | Q Q Q Q | 287, 288 287, 289 287, 290 287, 291 |



Table A7.1: Bromocarbons (C, H, O, N, Br) (...continued)

| Substance Formula (Trivial Name) [CAS Registry Number] InChIKey | $H_s^{cp}$ (at $T^\ominus$) $\left[\dfrac{\text{mol}}{\text{m}^3\,\text{Pa}}\right]$ | $\dfrac{\text{d}\ln H_s^{cp}}{\text{d}(1/T)}$ [K] | Reference | Type | Note |
|---|---|---|---|---|---|
| 4,5,6,7-tetrabromo-1,3-isobenzofurandione | $6.1\times10^1$ | | Zhang et al. (2010) | Q | 287, 288 |
| C$_8$Br$_4$O$_3$ | $4.4\times10^5$ | | Zhang et al. (2010) | Q | 287, 289 |
| [632-79-1] | $2.4\times10^2$ | | Zhang et al. (2010) | Q | 287, 290 |
| QHWKHLYUUZGSCW-UHFFFAOYSA-N | $8.0\times10^2$ | | Zhang et al. (2010) | Q | 287, 291 |
| allyl 2,4,6-tribromophenyl ether | $3.8\times10^{-1}$ | | Zhang et al. (2010) | Q | 287, 288 |
| C$_9$H$_7$Br$_3$O | $1.3\times10^{-1}$ | | Zhang et al. (2010) | Q | 287, 289 |
| [3278-89-5] | $2.0\times10^{-1}$ | | Zhang et al. (2010) | Q | 287, 290 |
| RZLLIOPGUFOWOD-UHFFFAOYSA-N | $6.2\times10^{-1}$ | | Zhang et al. (2010) | Q | 287, 291 |
| 2,4-dibromo-6-methylphenyl glycidyl ether | $8.2\times10^1$ | | Zhang et al. (2010) | Q | 287, 288 |
| C$_{10}$H$_{10}$Br$_2$O$_2$ | 7.0 | | Zhang et al. (2010) | Q | 287, 289 |
| [75150-13-9] | $5.2\times10^1$ | | Zhang et al. (2010) | Q | 287, 290 |
| XQTJZNGNEJLXTR-UHFFFAOYSA-N | 5.4 | | Zhang et al. (2010) | Q | 287, 291 |
| 2-(2,4,6-tribromophenoxy)ethyl acrylate | $2.9\times10^2$ | | Zhang et al. (2010) | Q | 287, 288 |
| C$_{11}$H$_9$Br$_3$O$_3$ | $1.6\times10^1$ | | Zhang et al. (2010) | Q | 287, 289 |
| [7347-19-5] | $4.3\times10^3$ | | Zhang et al. (2010) | Q | 287, 290 |
| AMBJXYFIMKHOQE-UHFFFAOYSA-N | $1.3\times10^2$ | | Zhang et al. (2010) | Q | 287, 291 |
| 2,3,7,8-tetrabromodibenzo-$p$-dioxin C$_{12}$H$_4$Br$_4$O$_2$ [50585-41-6] JZLQUWSWOJPCAK-UHFFFAOYSA-N | 3.0 | | Ebert et al. (2023) | ? | 318 |
| octabromodibenzo-$p$-dioxin C$_{12}$Br$_8$O$_2$ [2170-45-8] XAHTWKGGNHXJRP-UHFFFAOYSA-N | $5.1\times10^1$ | | Ebert et al. (2023) | ? | 318 |
| 4,4'-methylenebis(2,6-dibromophenol) | $7.5\times10^7$ | | Zhang et al. (2010) | Q | 287, 288 |
| C$_{13}$H$_8$Br$_4$O$_2$ | $9.0\times10^1$ | | Zhang et al. (2010) | Q | 287, 289 |
| [21825-03-6] | $1.3\times10^4$ | | Zhang et al. (2010) | Q | 287, 290 |
| WPZJSWWEEJJSIZ-UHFFFAOYSA-N | $3.4\times10^3$ | | Zhang et al. (2010) | Q | 287, 291 |
| 1,1'-[1,2-ethanediylbis(oxy)]bis pentabromobenzene | $5.3\times10^4$ | | Zhang et al. (2010) | Q | 287, 288 |
| C$_{14}$H$_4$Br$_{10}$O$_2$ | $8.6\times10^2$ | | Zhang et al. (2010) | Q | 287, 289 |
| [61262-53-1] | $2.2\times10^2$ | | Zhang et al. (2010) | Q | 287, 290 |
| JJEPQBZQAGCZTH-UHFFFAOYSA-N | $1.1\times10^3$ | | Zhang et al. (2010) | Q | 287, 291 |
| 4,4'-dibromobenzil | $8.0\times10^3$ | | Zhang et al. (2010) | Q | 287, 288 |
| C$_{14}$H$_8$Br$_2$O$_2$ | $2.6\times10^3$ | | Zhang et al. (2010) | Q | 287, 289 |
| [35578-47-3] | $1.9\times10^2$ | | Zhang et al. (2010) | Q | 287, 290 |
| NYCBYBDDECLFPE-UHFFFAOYSA-N | $1.3\times10^5$ | | Zhang et al. (2010) | Q | 287, 291 |



Table A7.1: Bromocarbons (C, H, O, N, Br) (... continued)

| Substance<br>Formula<br>(Trivial Name)<br>[CAS Registry Number]<br>InChIKey | $H_s^{cp}$<br>(at $T^\ominus$)<br>$\left[\dfrac{\text{mol}}{\text{m}^3\,\text{Pa}}\right]$ | $\dfrac{\text{d}\ln H_s^{cp}}{\text{d}(1/T)}$<br>[K] | Reference | Type | Note |
|---|---|---|---|---|---|
| 1,2-bis(2,4,6-tribromophenoxy)ethane | $1.8\times10^1$ | | Kuramochi et al. (2014) | V | |
| $C_{14}H_8Br_6O_2$ | $2.3\times10^1$ | | HSDB (2015) | Q | 99 |
| (BTBPE) | $6.4\times10^1$ | | Xiao et al. (2012) | Q | |
| [37853-59-1] | $1.3\times10^3$ | | Zhang et al. (2010) | Q | 287, 288 |
| YATIGPZCMOYEGE-UHFFFAOYSA-N | $7.3\times10^1$ | | Zhang et al. (2010) | Q | 287, 289 |
| | $1.1\times10^3$ | | Zhang et al. (2010) | Q | 287, 290 |
| | $1.0\times10^3$ | | Zhang et al. (2010) | Q | 287, 291 |
| 2-ethylhexyl-2,3,4,5-tetrabromobenzoate | 1.6 | | Xiao et al. (2012) | Q | |
| $C_{15}H_{18}Br_4O_2$ | | | | | |
| (EHTeBB) | | | | | |
| [183658-27-7] | | | | | |
| HVDXCGSGEQKWGB-UHFFFAOYSA-N | | | | | |
| tribromobisphenol A | $1.1\times10^2$ | | HSDB (2015) | Q | 447 |
| $C_{15}H_{13}Br_3O_2$ | | | | | |
| [6386-73-8] | | | | | |
| WYBOEVJIVYIEJL-UHFFFAOYSA-N | | | | | |
| 4,4'-(1-methylethylidene)bis(2,6-dibromophenol) | $2.4\times10^2$ | | HSDB (2015) | V | |
| $C_{15}H_{12}Br_4O_2$ | $4.2\times10^7$ | | Zhang et al. (2010) | Q | 287, 288 |
| [79-94-7] | $3.9\times10^1$ | | Zhang et al. (2010) | Q | 287, 289 |
| VEORPZCZECFIRK-UHFFFAOYSA-N | $8.0\times10^4$ | | Zhang et al. (2010) | Q | 287, 290 |
| | $1.6\times10^3$ | | Zhang et al. (2010) | Q | 287, 291 |
| 4-[2-[2,6-bis(bromanyl)-4-oxidanyl-phenyl]propan-2-yl]-3,5-bis(bromanyl)phenol | $4.2\times10^7$ | | Zhang et al. (2010) | Q | 287, 288 |
| $C_{15}H_{12}Br_4O_2$ | $2.0\times10^7$ | | Zhang et al. (2010) | Q | 287, 289 |
| [94334-64-2] | $2.2\times10^7$ | | Zhang et al. (2010) | Q | 287, 290 |
| KIZJVNGSIWXYTL-UHFFFAOYSA-N | $1.7\times10^8$ | | Zhang et al. (2010) | Q | 287, 291 |
| 2-(2-hydroxyethoxy)ethyl 2-hydroxypropyl 3,4,5,6-tetrabromophthalate | $3.6\times10^{10}$ | | Zhang et al. (2010) | Q | 287, 288 |
| $C_{15}H_{16}Br_4O_7$ | $1.5\times10^{11}$ | | Zhang et al. (2010) | Q | 287, 289 |
| [20566-35-2] | $3.1\times10^{13}$ | | Zhang et al. (2010) | Q | 287, 290 |
| OQHHASWHOGRCRC-UHFFFAOYSA-N | $5.7\times10^{10}$ | | Zhang et al. (2010) | Q | 287, 291 |
| bromopropylate | $2.1\times10^1$ | | Duchowicz et al. (2020) | V | 186 |
| $C_{17}H_{16}Br_2O_3$ | $2.1\times10^1$ | | HSDB (2015) | V | |
| [18181-80-1] | $4.2\times10^2$ | | Duchowicz et al. (2020) | Q | |
| FOANIXZHAMJWOI-UHFFFAOYSA-N | | | | | |



Table A7.1: Bromocarbons (C, H, O, N, Br) (... continued)

| Substance Formula (Trivial Name) [CAS Registry Number] InChIKey | $H_s^{cp}$ (at $T^\ominus$) $\left[\dfrac{\text{mol}}{\text{m}^3\,\text{Pa}}\right]$ | $\dfrac{\mathrm{d}\ln H_s^{cp}}{\mathrm{d}(1/T)}$ [K] | Reference | Type | Note |
|---|---|---|---|---|---|
| 1,2,4,5-tetrabromo-3,6-bis(pentabromophenoxy)benzene | $1.5\times10^6$ | | Zhang et al. (2010) | Q | 287, 288 |
| $C_{18}O_2Br_{14}$ | $4.1\times10^5$ | | Zhang et al. (2010) | Q | 287, 289 |
| [58965-66-5] | $2.1\times10^6$ | | Zhang et al. (2010) | Q | 287, 290 |
| YMIUHIAWWDYGGU-UHFFFAOYSA-N | $6.7\times10^5$ | | Zhang et al. (2010) | Q | 287, 291 |
| 2,2-bis(3,5-dibromo-4-(2-hydroxyethoxy)phenyl)propane | $5.6\times10^7$ | | Zhang et al. (2010) | Q | 287, 288 |
| $C_{19}H_{20}Br_4O_4$ | $1.5\times10^8$ | | Zhang et al. (2010) | Q | 287, 289 |
| [4162-45-2] | $6.1\times10^9$ | | Zhang et al. (2010) | Q | 287, 290 |
| RVHUMFJSCJBNGS-UHFFFAOYSA-N | $2.5\times10^8$ | | Zhang et al. (2010) | Q | 287, 291 |
| metrafenone $C_{19}H_{21}BrO_5$ [220899-03-6] AMSPWOYQQAWRRM-UHFFFAOYSA-N | 7.6 | | Maniere et al. (2011) | ? | 12, 165 |
| solvent red 43 | $4.4\times10^{12}$ | | Zhang et al. (2010) | Q | 287, 288 |
| $C_{20}H_8Br_4O_5$ | $1.5\times10^8$ | | Zhang et al. (2010) | Q | 287, 289 |
| [15086-94-9] | $2.7\times10^{10}$ | | Zhang et al. (2010) | Q | 287, 290 |
| DBZJJPROPLPMSN-UHFFFAOYSA-N | $2.9\times10^8$ | | Zhang et al. (2010) | Q | 287, 291 |
| 2,2-bis[4-(2,3-dibromopropoxy)-3,5-dibromophenyl]-propane | $2.4\times10^5$ | | Zhang et al. (2010) | Q | 287, 288 |
| $C_{21}H_{20}Br_8O_2$ | $4.0\times10^4$ | | Zhang et al. (2010) | Q | 287, 289 |
| [21850-44-2] | $1.7\times10^5$ | | Zhang et al. (2010) | Q | 287, 290 |
| LXIZRZRTWSDLKK-UHFFFAOYSA-N | $8.6\times10^4$ | | Zhang et al. (2010) | Q | 287, 291 |
| 2,2-bis(4-allyloxy-3,5-dibromophenyl)propane | $7.7\times10^1$ | | Zhang et al. (2010) | Q | 287, 288 |
| $C_{21}H_{20}Br_4O_2$ | $1.3\times10^1$ | | Zhang et al. (2010) | Q | 287, 289 |
| [25327-89-3] | $1.7\times10^2$ | | Zhang et al. (2010) | Q | 287, 290 |
| PWXTUWQHMIFLKL-UHFFFAOYSA-N | $1.9\times10^2$ | | Zhang et al. (2010) | Q | 287, 291 |
| AC1MJ2TG | $1.3\times10^8$ | | Zhang et al. (2010) | Q | 287, 288 |
| $C_{21}H_{24}Br_4O_4$ | $4.7\times10^6$ | | Zhang et al. (2010) | Q | 287, 289 |
| [33294-14-3] | $9.2\times10^6$ | | Zhang et al. (2010) | Q | 287, 290 |
| UTMWVHZWNKBNKF-UHFFFAOYSA-N | $1.2\times10^6$ | | Zhang et al. (2010) | Q | 287, 291 |
| tetrabromophenolphthalein, ethyl ester | $1.0\times10^{11}$ | | Zhang et al. (2010) | Q | 287, 288 |
| $C_{22}H_{14}Br_4O_4$ | $1.2\times10^7$ | | Zhang et al. (2010) | Q | 287, 289 |
| [1176-74-5] | $3.1\times10^{10}$ | | Zhang et al. (2010) | Q | 287, 290 |
| SQFXATUXPUCFFO-UHFFFAOYSA-N | $3.5\times10^8$ | | Zhang et al. (2010) | Q | 287, 291 |
| 4,10-dibromodibenzo[$def,mno$]chrysene-6,12-dione | $5.8\times10^6$ | | Zhang et al. (2010) | Q | 287, 288 |
| $C_{22}H_8Br_2O_2$ | $2.7\times10^5$ | | Zhang et al. (2010) | Q | 287, 289 |
| [4378-61-4] | $4.1\times10^6$ | | Zhang et al. (2010) | Q | 287, 290 |
| HTENFZMEHKCNMD-UHFFFAOYSA-N | $1.1\times10^8$ | | Zhang et al. (2010) | Q | 287, 291 |



Table A7.1: Bromocarbons (C, H, O, N, Br) (...continued)

| Substance<br>Formula<br>(Trivial Name)<br>[CAS Registry Number]<br>InChIKey | $H_s^{cp}$<br>(at $T^\ominus$)<br>$\left[\dfrac{\text{mol}}{\text{m}^3\,\text{Pa}}\right]$ | $\dfrac{\text{d}\ln H_s^{cp}}{\text{d}(1/T)}$<br><br>[K] | Reference | Type | Note |
|---|---|---|---|---|---|
| bis(2-ethylhexyl)-3,4,5,6-<br>tetrabromophthalate<br>$C_{24}H_{34}Br_4O_4$<br>(TBPH)<br>[26040-51-7]<br>UUEDINPOVKWVAZ-UHFFFAOYSA-N | $4.0\times10^2$ | | Xiao et al. (2012) | Q | |
| bromadiolone<br>$C_{30}H_{23}BrO_4$<br>[28772-56-7]<br>OWNRRUFOJXFKCU-UHFFFAOYSA-N | $1.1\times10^6$<br>$1.1\times10^6$ | | HSDB (2015)<br>Maniere et al. (2011) | V<br>? | <br>241, 165 |
| brodifacoum<br>$C_{31}H_{23}BrO_3$<br>[56073-10-0]<br>VEUZZDOCACZPRY-UHFFFAOYSA-N | $4.6\times10^2$ | | Rubbiani (2013) | ? | |



## A7.2 Polybrominated diphenyl ethers (PBDEs)

Table A7.2: Polybrominated diphenyl ethers (PBDEs)

| Substance Formula (Trivial Name) [CAS Registry Number] InChIKey | $H_s^{cp}$ (at $T^\ominus$) $\left[\dfrac{\text{mol}}{\text{m}^3\,\text{Pa}}\right]$ | $\dfrac{\text{d}\ln H_s^{cp}}{\text{d}(1/T)}$ [K] | Reference | Type | Note |
|---|---|---|---|---|---|
| 2-bromodiphenyl ether $C_{12}H_9BrO$ (PBDE-1) [36563-47-0] RRWFUWRLNIZICP-UHFFFAOYSA-N | $2.3\times10^{-2}$ | 7400 | Long et al. (2017) | Q | 287 |
| 3-bromodiphenyl ether $C_{12}H_9BrO$ (PBDE-2) [6876-00-2] AHDAKFFMKLQPTD-UHFFFAOYSA-N | $2.3\times10^{-2}$ | 7400 | Long et al. (2017) | Q | 287 |
| 4-bromodiphenyl ether $C_{12}H_9BrO$ (PBDE-3) [101-55-3] JDUYPUMQALQRCN-UHFFFAOYSA-N | $5.0\times10^{-2}$ $4.3\times10^{-2}$ $5.8\times10^{-2}$ $9.6\times10^{-2}$ $1.4\times10^{-2}$ $8.2\times10^{-2}$ | 5500 7400 | Lau et al. (2006) Lau et al. (2006) Charles and Destaillats (2005) Mackay et al. (1993) Long et al. (2017) HSDB (2015) | M M M V Q Q | 719 720 287 99 |
| 2,2'-dibromodiphenyl ether $C_{12}H_8Br_2O$ (PBDE-4) [51452-87-0] JMSKYMHFNWGUJG-UHFFFAOYSA-N | $9.5\times10^{-2}$ | 7400 | Long et al. (2017) | Q | 287 |
| 2,3-dibromodiphenyl ether $C_{12}H_8Br_2O$ (PBDE-5) [446254-14-4] JTYRXXKXOULVAP-UHFFFAOYSA-N | $5.1\times10^{-2}$ | 7400 | Long et al. (2017) | Q | 287 |
| 2,3'-dibromodiphenyl ether $C_{12}H_8Br_2O$ (PBDE-6) [147217-72-9] GODQTPRKFHOLPH-UHFFFAOYSA-N | $9.4\times10^{-2}$ | 7400 | Long et al. (2017) | Q | 287 |
| 2,4-dibromodiphenyl ether $C_{12}H_8Br_2O$ (PBDE-7) [171977-44-9] JMCIHKKTRDLVCO-UHFFFAOYSA-N | $6.9\times10^{-2}$ | 7400 | Long et al. (2017) | Q | 287 |



Table A7.2: Polybrominated diphenyl ethers (PBDEs) (. . . continued)

| Substance Formula (Trivial Name) [CAS Registry Number] InChIKey | $H_s^{cp}$ (at $T^{\ominus}$) $\left[\dfrac{\text{mol}}{\text{m}^3\,\text{Pa}}\right]$ | $\dfrac{\text{d}\ln H_s^{cp}}{\text{d}(1/T)}$ [K] | Reference | Type | Note |
|---|---|---|---|---|---|
| 2,4'-dibromodiphenyl ether $C_{12}H_8Br_2O$ (PBDE-8) [147217-71-8] RJQLQJZMLISKRJ-UHFFFAOYSA-N | $5.8\times10^{-2}$ | 7400 | Long et al. (2017) | Q | 287 |
| 2,5-dibromodiphenyl ether $C_{12}H_8Br_2O$ (PBDE-9) [337513-66-3] URDWJMUOJJSXAE-UHFFFAOYSA-N | $8.0\times10^{-2}$ | 7400 | Long et al. (2017) | Q | 287 |
| 2,6-dibromodiphenyl ether $C_{12}H_8Br_2O$ (PBDE-10) [51930-04-2] MUVDKHMQIZJFTC-UHFFFAOYSA-N | $8.3\times10^{-2}$ | 7400 | Long et al. (2017) | Q | 287 |
| 3,3'-dibromodiphenyl ether $C_{12}H_8Br_2O$ (PBDE-11) [6903-63-5] ALSVFJIXSNRBLE-UHFFFAOYSA-N | $7.8\times10^{-2}$ | 7400 | Long et al. (2017) | Q | 287 |
| 3,4-dibromodiphenyl ether $C_{12}H_8Br_2O$ (PBDE-12) [189084-59-1] SUUJFDKVPDCZQZ-UHFFFAOYSA-N | $3.3\times10^{-2}$ | 7400 | Long et al. (2017) | Q | 287 |
| 3,4'-dibromodiphenyl ether $C_{12}H_8Br_2O$ (PBDE-13) [83694-71-7] BGPOVBPKODCMMN-UHFFFAOYSA-N | $5.5\times10^{-2}$ | 7400 | Long et al. (2017) | Q | 287 |
| 3,5-dibromodiphenyl ether $C_{12}H_8Br_2O$ (PBDE-14) [46438-88-4] FOXBZJLXVUHYQZ-UHFFFAOYSA-N | $9.9\times10^{-2}$ | 7400 | Long et al. (2017) | Q | 287 |
| 4,4'-dibromodiphenyl ether $C_{12}H_8Br_2O$ (PBDE-15) [2050-47-7] YAWIAFUBXXPJMQ-UHFFFAOYSA-N | $8.3\times10^{-2}$ $7.1\times10^{-2}$ $7.3\times10^{-2}$ $4.8\times10^{-2}$ $2.4\times10^{-1}$ $3.9\times10^{-2}$ $9.0\times10^{-2}$ | 4500 7400 | Lau et al. (2006) Lau et al. (2006) Charles and Destaillats (2005) Tittlemier et al. (2002) Wania and Dugani (2003) Long et al. (2017) Hilal et al. (2008) | M M M V R Q Q | 719 720 287 |





Table A7.2: Polybrominated diphenyl ethers (PBDEs) (. . . continued)

| Substance Formula (Trivial Name) [CAS Registry Number] InChIKey | $H_s^{cp}$ (at $T^{\ominus}$) $\left[\dfrac{\text{mol}}{\text{m}^3\,\text{Pa}}\right]$ | $\dfrac{\text{d}\ln H_s^{cp}}{\text{d}(1/T)}$ [K] | Reference | Type | Note |
|---|---|---|---|---|---|
| 2,2',3-tribromodiphenyl ether $C_{12}H_7Br_3O$ (PBDE-16) [147217-74-1] VRNGWCVCSHJUEJ-UHFFFAOYSA-N | $2.2\times10^{-1}$ | 7400 | Long et al. (2017) | Q | 287 |
| 2,2',4-tribromodiphenyl ether $C_{12}H_7Br_3O$ (PBDE-17) [147217-75-2] VYBFILXLBMWOLI-UHFFFAOYSA-N | $2.3\times10^{-1}$ | 7400 | Long et al. (2017) | Q | 287 |
| 2,2',5-tribromodiphenyl ether $C_{12}H_7Br_3O$ (PBDE-18) [407606-55-7] FAZLXBWRNJAGSV-UHFFFAOYSA-N | $2.4\times10^{-1}$ | 7400 | Long et al. (2017) | Q | 287 |
| 2,2',6-tribromodiphenyl ether $C_{12}H_7Br_3O$ (PBDE-19) [147217-73-0] YDFQHBRKURQGAH-UHFFFAOYSA-N | $3.6\times10^{-1}$ | 7400 | Long et al. (2017) | Q | 287 |
| 2,3,3'-tribromodiphenyl ether $C_{12}H_7Br_3O$ (PBDE-20) [147217-76-3] RQJUBSPXDSGLRB-UHFFFAOYSA-N | $1.8\times10^{-1}$ | 7400 | Long et al. (2017) | Q | 287 |
| 2,3,4-tribromodiphenyl ether $C_{12}H_7Br_3O$ (PBDE-21) [337513-67-4] RXWRVYYPLRPDOS-UHFFFAOYSA-N | $1.0\times10^{-1}$ | 7400 | Long et al. (2017) | Q | 287 |
| 2,3,4'-tribromodiphenyl ether $C_{12}H_7Br_3O$ (PBDE-22) [446254-15-5] WZHNIFQVNBINLF-UHFFFAOYSA-N | $1.2\times10^{-1}$ | 7400 | Long et al. (2017) | Q | 287 |
| 2,3,5-tribromodiphenyl ether $C_{12}H_7Br_3O$ (PBDE-23) [446254-16-6] XQHLKDAUZRXBGC-UHFFFAOYSA-N | $2.3\times10^{-1}$ | 7400 | Long et al. (2017) | Q | 287 |



Table A7.2: Polybrominated diphenyl ethers (PBDEs) (. . . continued)

| Substance<br>Formula<br>(Trivial Name)<br>[CAS Registry Number]<br>InChIKey | $H_s^{cp}$<br>(at $T^{\ominus}$)<br><br>$\left[\dfrac{\text{mol}}{\text{m}^3\,\text{Pa}}\right]$ | $\dfrac{\text{d}\ln H_s^{cp}}{\text{d}(1/T)}$<br><br><br>[K] | Reference | Type | Note |
|---|---|---|---|---|---|
| 2,3,6-tribromodiphenyl ether<br>$C_{12}H_7Br_3O$<br>(PBDE-24)<br>[218304-36-0]<br>GFLRHBRMAZDOIG-UHFFFAOYSA-N | $1.8\times10^{-1}$ | 7400 | Long et al. (2017) | Q | 287 |
| 2,3',4-tribromodiphenyl ether<br>$C_{12}H_7Br_3O$<br>(PBDE-25)<br>[147217-77-4]<br>AURKEOPYVUYTLO-UHFFFAOYSA-N | $2.3\times10^{-1}$ | 7400 | Long et al. (2017) | Q | 287 |
| 2,3',5-tribromodiphenyl ether<br>$C_{12}H_7Br_3O$<br>(PBDE-26)<br>[337513-75-4]<br>VUOBKVBAFJQQDB-UHFFFAOYSA-N | $2.5\times10^{-1}$ | 7400 | Long et al. (2017) | Q | 287 |
| 2,3',6-tribromodiphenyl ether<br>$C_{12}H_7Br_3O$<br>(PBDE-27)<br>[337513-53-8]<br>JUPZALSVNWJHII-UHFFFAOYSA-N | $3.4\times10^{-1}$ | 7400 | Long et al. (2017) | Q | 287 |
| 2,4,4'-tribromodiphenyl ether<br>$C_{12}H_7Br_3O$<br>(PBDE-28)<br>[41318-75-6]<br>UPNBETHEXPIWQX-UHFFFAOYSA-N | $1.6\times10^{-1}$<br>$1.1\times10^{-1}$<br>$7.7\times10^{-2}$<br>$1.8\times10^{-1}$<br>$1.2\times10^{-1}$<br>$2.0\times10^{-1}$<br>$5.2\times10^{-1}$<br>$1.8\times10^{-1}$<br>$1.4\times10^{-1}$ | 8100<br><br><br>7400<br>12000<br><br><br>7400<br> | Long et al. (2017)<br>Lau et al. (2006)<br>Lau et al. (2006)<br>Cetin and Odabasi (2005)<br>Charles and Destaillats (2005)<br>Tittlemier et al. (2002)<br>Wania and Dugani (2003)<br>Long et al. (2017)<br>Hilal et al. (2008) | M<br>M<br>M<br>M<br>M<br>V<br>R<br>Q<br>Q | 287<br>719<br>720<br><br>33<br><br><br>287<br> |
| 2,4,5-tribromodiphenyl ether<br>$C_{12}H_7Br_3O$<br>(PBDE-29)<br>[337513-56-1]<br>LTMKAFUXYKEDLR-UHFFFAOYSA-N | $1.6\times10^{-1}$ | 7400 | Long et al. (2017) | Q | 287 |
| 2,4,6-tribromodiphenyl ether<br>$C_{12}H_7Br_3O$<br>(PBDE-30)<br>[155999-95-4]<br>TVZAPPGLBLTACB-UHFFFAOYSA-N | $3.4\times10^{-1}$ | 7400 | Long et al. (2017) | Q | 287 |





Table A7.2: Polybrominated diphenyl ethers (PBDEs) (...continued)

| Substance<br>Formula<br>(Trivial Name)<br>[CAS Registry Number]<br>InChIKey | $H_s^{cp}$<br>(at $T^{\ominus}$)<br><br>$\left[\dfrac{\mathrm{mol}}{\mathrm{m^3\,Pa}}\right]$ | $\dfrac{\mathrm{d}\ln H_s^{cp}}{\mathrm{d}(1/T)}$<br><br>[K] | Reference | Type | Note |
|---|---|---|---|---|---|
| 2,4',5-tribromodiphenyl ether<br>$C_{12}H_7Br_3O$<br>(PBDE-31)<br>[65075-08-3]<br>PURZBWMLFRWRMG-UHFFFAOYSA-N | $1.9\times10^{-1}$ | 7400 | Long et al. (2017) | Q | 287 |
| 2,4',6-tribromodiphenyl ether<br>$C_{12}H_7Br_3O$<br>(PBDE-32)<br>[189084-60-4]<br>TYDVYKIQSZGUMV-UHFFFAOYSA-N | $1.6\times10^{-1}$ | 7400 | Long et al. (2017) | Q | 287 |
| 2,3',4'-tribromodiphenyl ether<br>$C_{12}H_7Br_3O$<br>(PBDE-33)<br>[49690-94-0]<br>BUQBQEYUVAKJQK-UHFFFAOYSA-N | $1.3\times10^{-1}$ | 7400 | Long et al. (2017) | Q | 287 |
| 2,3',5'-tribromodiphenyl ether<br>$C_{12}H_7Br_3O$<br>(PBDE-34)<br>[446254-17-7]<br>XMNXHCHZIPYCNA-UHFFFAOYSA-N | $4.2\times10^{-1}$ | 7400 | Long et al. (2017) | Q | 287 |
| 3,3',4-tribromodiphenyl ether<br>$C_{12}H_7Br_3O$<br>(PBDE-35)<br>[147217-80-9]<br>CDVYKQPKJYPWRO-UHFFFAOYSA-N | $1.2\times10^{-1}$ | 7400 | Long et al. (2017) | Q | 287 |
| 3,3',5-tribromodiphenyl ether<br>$C_{12}H_7Br_3O$<br>(PBDE-36)<br>[147217-79-6]<br>XUKPJLVONRTECE-UHFFFAOYSA-N | $2.8\times10^{-1}$ | 7400 | Long et al. (2017) | Q | 287 |
| 3,4,4'-tribromodiphenyl ether<br>$C_{12}H_7Br_3O$<br>(PBDE-37)<br>[147217-81-0]<br>YALAYFVVZFORPV-UHFFFAOYSA-N | $9.0\times10^{-2}$ | 7400 | Long et al. (2017) | Q | 287 |
| 3,4,5-tribromodiphenyl ether<br>$C_{12}H_7Br_3O$<br>(PBDE-38)<br>[337513-54-9]<br>DPGVQKLGQZZLMI-UHFFFAOYSA-N | $9.9\times10^{-2}$ | 7400 | Long et al. (2017) | Q | 287 |



Table A7.2: Polybrominated diphenyl ethers (PBDEs) (...continued)

| Substance<br>Formula<br>(Trivial Name)<br>[CAS Registry Number]<br>InChIKey | $H_s^{cp}$<br>(at $T^\ominus$)<br>$\left[\dfrac{\mathrm{mol}}{\mathrm{m^3\,Pa}}\right]$ | $\dfrac{\mathrm{d}\ln H_s^{cp}}{\mathrm{d}(1/T)}$<br><br>[K] | Reference | Type | Note |
|---|---|---|---|---|---|
| 3,4',5-tribromodiphenyl ether<br>$C_{12}H_7Br_3O$<br>(PBDE-39)<br>[407606-57-9]<br>UFFNOPDHJNQYKD-UHFFFAOYSA-N | $2.0\times10^{-1}$ | 7400 | Long et al. (2017) | Q | 287 |
| 2,2',3,3'-tetrabromodiphenyl ether<br>$C_{12}H_6Br_4O$<br>(PBDE-40)<br>[337513-77-6]<br>SXSUUFZWSVMTRL-UHFFFAOYSA-N | $4.8\times10^{-1}$ | 7400 | Long et al. (2017) | Q | 287 |
| 2,2',3,4-tetrabromodiphenyl ether<br>$C_{12}H_6Br_4O$<br>(PBDE-41)<br>[337513-68-5]<br>UAEBSKBXZAIRMX-UHFFFAOYSA-N | $3.2\times10^{-1}$ | 7400 | Long et al. (2017) | Q | 287 |
| 2,2',3,4'-tetrabromodiphenyl ether<br>$C_{12}H_6Br_4O$<br>(PBDE-42)<br>[446254-18-8]<br>HQDQKPAHIDGGMH-UHFFFAOYSA-N | $5.2\times10^{-1}$ | 7400 | Long et al. (2017) | Q | 287 |
| 2,2',3,5-tetrabromodiphenyl ether<br>$C_{12}H_6Br_4O$<br>(PBDE-43)<br>[446254-19-9]<br>LKMQHSYDVDIECC-UHFFFAOYSA-N | $6.6\times10^{-1}$ | 7400 | Long et al. (2017) | Q | 287 |
| 2,2',3,5'-tetrabromodiphenyl ether<br>$C_{12}H_6Br_4O$<br>(PBDE-44)<br>[446254-20-2]<br>VBGBGTYMDIVKNK-UHFFFAOYSA-N | $5.7\times10^{-1}$ | 7400 | Long et al. (2017) | Q | 287 |
| 2,2',3,6-tetrabromodiphenyl ether<br>$C_{12}H_6Br_4O$<br>(PBDE-45)<br>[446254-21-3]<br>VTFWUBIOZQCMQS-UHFFFAOYSA-N | $7.3\times10^{-1}$ | 7400 | Long et al. (2017) | Q | 287 |
| 2,2',3,6'-tetrabromodiphenyl ether<br>$C_{12}H_6Br_4O$<br>(PBDE-46)<br>[446254-22-4]<br>GBUUKJRFSKCMTB-UHFFFAOYSA-N | 1.1 | 7400 | Long et al. (2017) | Q | 287 |



Table A7.2: Polybrominated diphenyl ethers (PBDEs) (...continued)

| Substance<br>Formula<br>(Trivial Name)<br>[CAS Registry Number]<br>InChIKey | $H_s^{cp}$ (at $T^{\ominus}$) $\left[\dfrac{\mathrm{mol}}{\mathrm{m^3\,Pa}}\right]$ | $\dfrac{\mathrm{d}\ln H_s^{cp}}{\mathrm{d}(1/T)}$ [K] | Reference | Type | Note |
|---|---|---|---|---|---|
| 2,2',4,4'-tetrabromodiphenyl ether<br>$C_{12}H_6Br_4O$<br>(PBDE-47)<br>[5436-43-1]<br>XYBSIYMGXVUVGY-UHFFFAOYSA-N | $9.1\times10^{-1}$<br>$1.6\times10^{-1}$<br>$1.7\times10^{-1}$<br>$8.7\times10^{-1}$<br>$1.7\times10^{-1}$<br>$9.3\times10^{-1}$<br>$6.7\times10^{-1}$<br>$9.0\times10^{-1}$<br>$8.5\times10^{-1}$<br>$2.2\times10^{-1}$ | 7400<br><br><br>7300<br>620<br><br><br><br>7400 | Long et al. (2017)<br>Lau et al. (2006)<br>Lau et al. (2006)<br>Cetin and Odabasi (2005)<br>Charles and Destaillats (2005)<br>Kuramochi et al. (2014)<br>Tittlemier et al. (2002)<br>Wania and Dugani (2003)<br>Long et al. (2017)<br>Hilal et al. (2008) | M<br>M<br>M<br>M<br>M<br>V<br>V<br>R<br>Q<br>Q | 287<br>719<br>720<br><br>42<br><br><br><br>287<br> |
| 2,2',4,5-tetrabromodiphenyl ether<br>$C_{12}H_6Br_4O$<br>(PBDE-48)<br>[337513-55-0]<br>FJGDNHOVDFREMP-UHFFFAOYSA-N | $4.8\times10^{-1}$ | 7400 | Long et al. (2017) | Q | 287 |
| 2,2',4,5'-tetrabromodiphenyl ether<br>$C_{12}H_6Br_4O$<br>(PBDE-49)<br>[243982-82-3]<br>QWVDUBDYUPHNHY-UHFFFAOYSA-N | $8.6\times10^{-1}$ | 7400 | Long et al. (2017) | Q | 287 |
| 2,2',4,6-tetrabromodiphenyl ether<br>$C_{12}H_6Br_4O$<br>(PBDE-50)<br>[446254-23-5]<br>FXUAKFRJBKFDSY-UHFFFAOYSA-N | 1.1 | 7400 | Long et al. (2017) | Q | 287 |
| 2,2',4,6'-tetrabromodiphenyl ether<br>$C_{12}H_6Br_4O$<br>(PBDE-51)<br>[189084-57-9]<br>WKBBBTLDLKYGBI-UHFFFAOYSA-N | 1.3 | 7400 | Long et al. (2017) | Q | 287 |
| 2,2',5,5'-tetrabromodiphenyl ether<br>$C_{12}H_6Br_4O$<br>(PBDE-52)<br>[446254-24-6]<br>CDTHXJORUCZHMD-UHFFFAOYSA-N | $8.9\times10^{-1}$ | 7400 | Long et al. (2017) | Q | 287 |
| 2,2',5,6'-tetrabromodiphenyl ether<br>$C_{12}H_6Br_4O$<br>(PBDE-53)<br>[446254-25-7]<br>SDVQGIMOFXMKHR-UHFFFAOYSA-N | 1.4 | 7400 | Long et al. (2017) | Q | 287 |



Table A7.2: Polybrominated diphenyl ethers (PBDEs) (...continued)

| Substance Formula (Trivial Name) [CAS Registry Number] InChIKey | $H_s^{cp}$ (at $T^{\ominus}$) $\left[\dfrac{\text{mol}}{\text{m}^3\,\text{Pa}}\right]$ | $\dfrac{\text{d}\ln H_s^{cp}}{\text{d}(1/T)}$ [K] | Reference | Type | Note |
|---|---|---|---|---|---|
| 2,2',6,6'-tetrabromodiphenyl ether $C_{12}H_6Br_4O$ (PBDE-54) [446254-26-8] WCDCHQGVTZHVSO-UHFFFAOYSA-N | 1.7 | 7400 | Long et al. (2017) | Q | 287 |
| 2,3,3',4-tetrabromodiphenyl ether $C_{12}H_6Br_4O$ (PBDE-55) [446254-27-9] VIHUMJGEWQPWOT-UHFFFAOYSA-N | $3.0\times10^{-1}$ | 7400 | Long et al. (2017) | Q | 287 |
| 2,3,3',4'-tetrabromodiphenyl ether $C_{12}H_6Br_4O$ (PBDE-56) [446254-28-0] NFOIVCGFYJIYIB-UHFFFAOYSA-N | $2.8\times10^{-1}$ | 7400 | Long et al. (2017) | Q | 287 |
| 2,3,3',5-tetrabromodiphenyl ether $C_{12}H_6Br_4O$ (PBDE-57) [337513-82-3] CSIFWDKYUJLQEB-UHFFFAOYSA-N | $6.2\times10^{-1}$ | 7400 | Long et al. (2017) | Q | 287 |
| 2,3,3',5'-tetrabromodiphenyl ether $C_{12}H_6Br_4O$ (PBDE-58) [446254-29-1] SWOYBZHGPZIRHS-UHFFFAOYSA-N | 1.3 | 7400 | Long et al. (2017) | Q | 287 |
| 2,3,3',6-tetrabromodiphenyl ether $C_{12}H_6Br_4O$ (PBDE-59) [446254-30-4] DMAMJZQQOWYEHT-UHFFFAOYSA-N | $6.5\times10^{-1}$ | 7400 | Long et al. (2017) | Q | 287 |
| 2,3,4,4'-tetrabromodiphenyl ether $C_{12}H_6Br_4O$ (PBDE-60) [446254-31-5] ARERIMFZYPFJAV-UHFFFAOYSA-N | $2.3\times10^{-1}$ | 7400 | Long et al. (2017) | Q | 287 |
| 2,3,4,5-tetrabromodiphenyl ether $C_{12}H_6Br_4O$ (PBDE-61) [446254-32-6] NDRSXNBQWAOQPP-UHFFFAOYSA-N | $3.2\times10^{-1}$ | 7400 | Long et al. (2017) | Q | 287 |



Table A7.2: Polybrominated diphenyl ethers (PBDEs) (...continued)

| Substance<br>Formula<br>(Trivial Name)<br>[CAS Registry Number]<br>InChIKey | $H_s^{cp}$<br>(at $T^{\ominus}$)<br>$\left[\dfrac{\text{mol}}{\text{m}^3\,\text{Pa}}\right]$ | $\dfrac{\text{d}\ln H_s^{cp}}{\text{d}(1/T)}$<br><br>[K] | Reference | Type | Note |
|---|---|---|---|---|---|
| 2,3,4,6-tetrabromodiphenyl ether<br>$C_{12}H_6Br_4O$<br>(PBDE-62)<br>[446254-33-7]<br>YIQYWYZZLOZVRM-UHFFFAOYSA-N | $5.5\times10^{-1}$ | 7400 | Long et al. (2017) | Q | 287 |
| 2,3,4',5-tetrabromodiphenyl ether<br>$C_{12}H_6Br_4O$<br>(PBDE-63)<br>[446254-34-8]<br>HNICYXFGCWPYGC-UHFFFAOYSA-N | $5.0\times10^{-1}$ | 7400 | Long et al. (2017) | Q | 287 |
| 2,3,4',6-tetrabromodiphenyl ether<br>$C_{12}H_6Br_4O$<br>(PBDE-64)<br>[446254-35-9]<br>LDCXVFJUWKKBNY-UHFFFAOYSA-N | $3.3\times10^{-1}$ | 7400 | Long et al. (2017) | Q | 287 |
| 2,3,5,6-tetrabromodiphenyl ether<br>$C_{12}H_6Br_4O$<br>(PBDE-65)<br>[446254-36-0]<br>HPEUYVBOPJQVPN-UHFFFAOYSA-N | $4.8\times10^{-1}$ | 7400 | Long et al. (2017) | Q | 287 |
| 2,3',4,4'-tetrabromodiphenyl ether<br>$C_{12}H_6Br_4O$<br>(PBDE-66)<br>[189084-61-5]<br>DHUMTYRHKMCVAG-UHFFFAOYSA-N | 2.0<br>$4.7\times10^{-1}$ | <br>7400 | Tittlemier et al. (2002)<br>Long et al. (2017) | V<br>Q | <br>287 |
| 2,3',4,5-tetrabromodiphenyl ether<br>$C_{12}H_6Br_4O$<br>(PBDE-67)<br>[446254-37-1]<br>OARGWSONVLGXQA-UHFFFAOYSA-N | $4.7\times10^{-1}$ | 7400 | Long et al. (2017) | Q | 287 |
| 2,3',4,5'-tetrabromodiphenyl ether<br>$C_{12}H_6Br_4O$<br>(PBDE-68)<br>[446254-38-2]<br>UFWGRLCUOLLWAO-UHFFFAOYSA-N | 1.6 | 7400 | Long et al. (2017) | Q | 287 |
| 2,3',4,6-tetrabromodiphenyl ether<br>$C_{12}H_6Br_4O$<br>(PBDE-69)<br>[327185-09-1]<br>NHZNRCYNZJADTG-UHFFFAOYSA-N | $9.8\times10^{-1}$ | 7400 | Long et al. (2017) | Q | 287 |



Table A7.2: Polybrominated diphenyl ethers (PBDEs) (...continued)

| Substance Formula (Trivial Name) [CAS Registry Number] InChIKey | $H_s^{cp}$ (at $T^\ominus$) $\left[\dfrac{\mathrm{mol}}{\mathrm{m^3\,Pa}}\right]$ | $\dfrac{\mathrm{d}\ln H_s^{cp}}{\mathrm{d}(1/T)}$ [K] | Reference | Type | Note |
|---|---|---|---|---|---|
| 2,3',4',5-tetrabromodiphenyl ether $C_{12}H_6Br_4O$ (PBDE-70) [446254-39-3] GHQMTYWQVJZWAR-UHFFFAOYSA-N | $5.2\times10^{-1}$ | 7400 | Long et al. (2017) | Q | 287 |
| 2,3',4',6-tetrabromodiphenyl ether $C_{12}H_6Br_4O$ (PBDE-71) [189084-62-6] COPAGYRSCJVION-UHFFFAOYSA-N | $5.2\times10^{-1}$ | 7400 | Long et al. (2017) | Q | 287 |
| 2,3',5,5'-tetrabromodiphenyl ether $C_{12}H_6Br_4O$ (PBDE-72) [446254-40-6] GBBNZKQTOOZGIS-UHFFFAOYSA-N | $9.6\times10^{-1}$ | 7400 | Long et al. (2017) | Q | 287 |
| 2,3',5',6-tetrabromodiphenyl ether $C_{12}H_6Br_4O$ (PBDE-73) [446254-41-7] WQFLVWXBCRJAQN-UHFFFAOYSA-N | 1.7 | 7400 | Long et al. (2017) | Q | 287 |
| 2,4,4',5-tetrabromodiphenyl ether $C_{12}H_6Br_4O$ (PBDE-74) [446254-42-8] LXCFDVVDUVPAGR-UHFFFAOYSA-N | $3.9\times10^{-1}$ | 7400 | Long et al. (2017) | Q | 287 |
| 2,4,4',6-tetrabromodiphenyl ether $C_{12}H_6Br_4O$ (PBDE-75) [189084-63-7] BWCNKMFFUGBFGB-UHFFFAOYSA-N | $6.3\times10^{-1}$ | 7400 | Long et al. (2017) | Q | 287 |
| 2,3',4',5'-tetrabromodiphenyl ether $C_{12}H_6Br_4O$ (PBDE-76) [446254-43-9] NCSWBJSFVPJPPK-UHFFFAOYSA-N | $4.7\times10^{-1}$ | 7400 | Long et al. (2017) | Q | 287 |
| 3,3',4,4'-tetrabromodiphenyl ether $C_{12}H_6Br_4O$ (PBDE-77) [93703-48-1] RYGLOWMCGZHYRQ-UHFFFAOYSA-N | $8.3\times10^{-1}$ $2.3\times10^{-1}$ | 7400 | Tittlemier et al. (2002) Long et al. (2017) | V Q | 287 |



Table A7.2: Polybrominated diphenyl ethers (PBDEs) (...continued)

| Substance Formula (Trivial Name) [CAS Registry Number] InChIKey | $H_s^{cp}$ (at $T^{\ominus}$) $\left[\dfrac{\text{mol}}{\text{m}^3\,\text{Pa}}\right]$ | $\dfrac{\mathrm{d}\ln H_s^{cp}}{\mathrm{d}(1/T)}$ [K] | Reference | Type | Note |
|---|---|---|---|---|---|
| 3,3',4,5-tetrabromodiphenyl ether $C_{12}H_6Br_4O$ (PBDE-78) [446254-45-1] HWOBLTZZSVXBOJ-UHFFFAOYSA-N | $2.9\times10^{-1}$ | 7400 | Long et al. (2017) | Q | 287 |
| 3,3',4,5'-tetrabromodiphenyl ether $C_{12}H_6Br_4O$ (PBDE-79) [446254-48-4] LELQGHJEUVRPEV-UHFFFAOYSA-N | $4.7\times10^{-1}$ | 7400 | Long et al. (2017) | Q | 287 |
| 3,3',5,5'-tetrabromodiphenyl ether $C_{12}H_6Br_4O$ (PBDE-80) [103173-66-6] HFIOZJQRZKNPKJ-UHFFFAOYSA-N | 1.1 | 7400 | Long et al. (2017) | Q | 287 |
| 3,4,4',5-tetrabromodiphenyl ether $C_{12}H_6Br_4O$ (PBDE-81) [446254-50-8] ULFOIXCXIWHJDS-UHFFFAOYSA-N | $2.4\times10^{-1}$ | 7400 | Long et al. (2017) | Q | 287 |
| 2,2',3,3',4-pentabromodiphenyl ether $C_{12}H_5Br_5O$ (PBDE-82) [327185-11-5] RQMSPGJESCCPQX-UHFFFAOYSA-N | $7.1\times10^{-1}$ | 7400 | Long et al. (2017) | Q | 287 |
| 2,2',3,3',5-pentabromodiphenyl ether $C_{12}H_5Br_5O$ (PBDE-83) [446254-51-9] XAHYSNUYJLNDBX-UHFFFAOYSA-N | 1.4 | 7400 | Long et al. (2017) | Q | 287 |
| 2,2',3,3',6-pentabromodiphenyl ether $C_{12}H_5Br_5O$ (PBDE-84) [446254-52-0] PPIZNRAVQHNLJM-UHFFFAOYSA-N | 1.8 | 7400 | Long et al. (2017) | Q | 287 |



Table A7.2: Polybrominated diphenyl ethers (PBDEs) (. . . continued)

| Substance<br>Formula<br>(Trivial Name)<br>[CAS Registry Number]<br>InChIKey | $H_s^{cp}$<br>(at $T^{\ominus}$)<br>$\left[\dfrac{\text{mol}}{\text{m}^3\,\text{Pa}}\right]$ | $\dfrac{\text{d}\ln H_s^{cp}}{\text{d}(1/T)}$<br><br>[K] | Reference | Type | Note |
|---|---|---|---|---|---|
| 2,2',3,4,4'-pentabromodiphenyl ether<br>$C_{12}H_5Br_5O$<br>(PBDE-85)<br>[182346-21-0]<br>DMLQSUZPTTUUDP-UHFFFAOYSA-N | 9.1<br><br>1.3 | <br><br>7400 | Tittlemier et al. (2002)<br><br>Long et al. (2017) | V<br><br>Q | <br><br>287 |
| 2,2',3,4,5-pentabromodiphenyl ether<br>$C_{12}H_5Br_5O$<br>(PBDE-86)<br>[446254-53-1]<br>YMVWYUWOUOQCQP-UHFFFAOYSA-N | $9.2\times10^{-1}$ | 7400 | Long et al. (2017) | Q | 287 |
| 2,2',3,4,5'-pentabromodiphenyl ether<br>$C_{12}H_5Br_5O$<br>(PBDE-87)<br>[446254-54-2]<br>WKYQUGCIKNOXFW-UHFFFAOYSA-N | $1.1\times10^{2}$ | 7400 | Long et al. (2017) | Q | 287 |
| 2,2',3,4,6-pentabromodiphenyl ether<br>$C_{12}H_5Br_5O$<br>(PBDE-88)<br>[446254-55-3]<br>OPZUHBCVIZNZFB-UHFFFAOYSA-N | 1.2 | 7400 | Long et al. (2017) | Q | 287 |
| 2,2',3,4,6'-pentabromodiphenyl ether<br>$C_{12}H_5Br_5O$<br>(PBDE-89)<br>[446254-56-4]<br>XGFLJLJXVIMCNR-UHFFFAOYSA-N | 1.8 | 7400 | Long et al. (2017) | Q | 287 |
| 2,2',3,4',5-pentabromodiphenyl ether<br>$C_{12}H_5Br_5O$<br>(PBDE-90)<br>[446254-57-5]<br>BATFXMGTVIESIQ-UHFFFAOYSA-N | 2.3 | 7400 | Long et al. (2017) | Q | 287 |
| 2,2',3,4',6-pentabromodiphenyl ether<br>$C_{12}H_5Br_5O$<br>(PBDE-91)<br>[446254-58-6]<br>HWNJTZKDPNZUSO-UHFFFAOYSA-N | 1.7 | 7400 | Long et al. (2017) | Q | 287 |



Table A7.2: Polybrominated diphenyl ethers (PBDEs) (…continued)

| Substance Formula (Trivial Name) [CAS Registry Number] InChIKey | $H_s^{cp}$ (at $T^{\ominus}$) $\left[\dfrac{\text{mol}}{\text{m}^3\,\text{Pa}}\right]$ | $\dfrac{\text{d}\ln H_s^{cp}}{\text{d}(1/T)}$ [K] | Reference | Type | Note |
|---|---|---|---|---|---|
| 2,2',3,5,5'-pentabromodiphenyl ether $C_{12}H_5Br_5O$ (PBDE-92) [446254-59-7] QWSQOVAGRDRZLM-UHFFFAOYSA-N | 1.9 | 7400 | Long et al. (2017) | Q | 287 |
| 2,2',3,5,6-pentabromodiphenyl ether $C_{12}H_5Br_5O$ (PBDE-93) [446254-60-0] BRTPVPJQMWLDNO-UHFFFAOYSA-N | 1.5 | 7400 | Long et al. (2017) | Q | 287 |
| 2,2',3,5,6'-pentabromodiphenyl ether $C_{12}H_5Br_5O$ (PBDE-94) [446254-61-1] JOPASNJHCFYVHD-UHFFFAOYSA-N | 3.8 | 7400 | Long et al. (2017) | Q | 287 |
| 2,2',3,5',6-pentabromodiphenyl ether $C_{12}H_5Br_5O$ (PBDE-95) [446254-62-2] BZDYRALIEYVMEP-UHFFFAOYSA-N | 1.8 | 7400 | Long et al. (2017) | Q | 287 |
| 2,2',3,6,6'-pentabromodiphenyl ether $C_{12}H_5Br_5O$ (PBDE-96) [446254-63-3] ZFCJNRDWGBZUED-UHFFFAOYSA-N | 3.9 | 7400 | Long et al. (2017) | Q | 287 |
| 2,2',3,4',5'-pentabromodiphenyl ether $C_{12}H_5Br_5O$ (PBDE-97) [446254-64-4] MAGYDGJRSCULJL-UHFFFAOYSA-N | 1.0 | 7400 | Long et al. (2017) | Q | 287 |
| 2,2',3,4',6'-pentabromodiphenyl ether $C_{12}H_5Br_5O$ (PBDE-98) [38463-82-0] OCLWEJVGAUFXQU-UHFFFAOYSA-N | 3.5 | 7400 | Long et al. (2017) | Q | 287 |



Table A7.2: Polybrominated diphenyl ethers (PBDEs) (. . . continued)

| Substance Formula (Trivial Name) [CAS Registry Number] InChIKey | $H_s^{cp}$ (at $T^\ominus$) $\left[\dfrac{\text{mol}}{\text{m}^3\,\text{Pa}}\right]$ | $\dfrac{\text{d}\ln H_s^{cp}}{\text{d}(1/T)}$ [K] | Reference | Type | Note |
|---|---|---|---|---|---|
| 2,2',4,4',5-pentabromodiphenyl ether | 1.5 | 8900 | Long et al. (2017) | M | 287 |
| $C_{12}H_5Br_5O$ | $6.2\times10^{-1}$ | | Lau et al. (2006) | M | 719 |
| (PBDE-99) | $3.3\times10^{-1}$ | | Lau et al. (2006) | M | 720 |
| [60348-60-9] | 1.5 | 8800 | Cetin and Odabasi (2005) | M | |
| WHPVYXDFIXRKLN-UHFFFAOYSA-N | $2.7\times10^{-1}$ | -6700 | Charles and Destaillats (2005) | M | 42 |
| | 2.1 | | Kuramochi et al. (2014) | V | |
| | 4.3 | | Tittlemier et al. (2002) | V | |
| | 1.9 | | Wania and Dugani (2003) | R | |
| | 1.6 | 7400 | Long et al. (2017) | Q | 287 |
| | 8.4 | | Zhang et al. (2010) | Q | 287, 288 |
| | 3.7 | | Zhang et al. (2010) | Q | 287, 289 |
| | $1.2\times10^2$ | | Zhang et al. (2010) | Q | 287, 290 |
| | $2.4\times10^1$ | | Zhang et al. (2010) | Q | 287, 291 |
| | $4.3\times10^{-1}$ | | Hilal et al. (2008) | Q | |
| 2,2',4,4',6-pentabromodiphenyl ether | 3.6 | 7300 | Long et al. (2017) | M | 287 |
| $C_{12}H_5Br_5O$ | $3.3\times10^{-1}$ | | Lau et al. (2006) | M | 719 |
| (PBDE-100) | $3.2\times10^{-1}$ | | Lau et al. (2006) | M | 720 |
| [189084-64-8] | 3.8 | 6800 | Cetin and Odabasi (2005) | M | |
| NSKIRYMHNFTRLR-UHFFFAOYSA-N | $1.9\times10^{-1}$ | 12 | Charles and Destaillats (2005) | M | 42 |
| | $1.4\times10^1$ | | Tittlemier et al. (2002) | V | |
| | 2.6 | | Wania and Dugani (2003) | R | |
| | 3.9 | 7400 | Long et al. (2017) | Q | 287 |
| | $3.7\times10^{-1}$ | | Hilal et al. (2008) | Q | |
| 2,2',4,5,5'-pentabromodiphenyl ether $C_{12}H_5Br_5O$ (PBDE-101) [446254-65-5] QUZWDWNIWWAQDI-UHFFFAOYSA-N | 1.4 | 7400 | Long et al. (2017) | Q | 287 |
| 2,2',4,5,6'-pentabromodiphenyl ether $C_{12}H_5Br_5O$ (PBDE-102) [446254-66-6] JHFMCUVMAIQWRI-UHFFFAOYSA-N | 2.8 | 7400 | Long et al. (2017) | Q | 287 |
| 2,2',4,5',6-pentabromodiphenyl ether $C_{12}H_5Br_5O$ (PBDE-103) [446254-67-7] RJEMKRNASVHYKR-UHFFFAOYSA-N | 3.4 | 7400 | Long et al. (2017) | Q | 287 |



Table A7.2: Polybrominated diphenyl ethers (PBDEs) (...continued)

| Substance Formula (Trivial Name) [CAS Registry Number] InChIKey | $H_s^{cp}$ (at $T^\ominus$) $\left[\dfrac{\text{mol}}{\text{m}^3\,\text{Pa}}\right]$ | $\dfrac{\text{d}\ln H_s^{cp}}{\text{d}(1/T)}$ [K] | Reference | Type | Note |
|---|---|---|---|---|---|
| 2,2',4,6,6'-pentabromodiphenyl ether C$_{12}$H$_5$Br$_5$O (PBDE-104) [446254-68-8] CRSCWEYUPUKHPI-UHFFFAOYSA-N | 6.7 | 7400 | Long et al. (2017) | Q | 287 |
| 2,3,3',4,4'-pentabromodiphenyl ether C$_{12}$H$_5$Br$_5$O (PBDE-105) [373594-78-6] LBPWAGZGYNOKAM-UHFFFAOYSA-N | $6.1\times10^{-1}$ | 7400 | Long et al. (2017) | Q | 287 |
| 2,3,3',4,5-pentabromodiphenyl ether C$_{12}$H$_5$Br$_5$O (PBDE-106) [446254-69-9] KLQKWMYXEWUAFP-UHFFFAOYSA-N | $8.5\times10^{-1}$ | 7400 | Long et al. (2017) | Q | 287 |
| 2,3,3',4',5-pentabromodiphenyl ether C$_{12}$H$_5$Br$_5$O (PBDE-107) [446254-70-2] OMGVAMFMRSETEG-UHFFFAOYSA-N | 1.9 | 7400 | Long et al. (2017) | Q | 287 |
| 2,3,3',4,5'-pentabromodiphenyl ether C$_{12}$H$_5$Br$_5$O (PBDE-108) [446254-71-3] VBKPKHVLHGOKOJ-UHFFFAOYSA-N | 2.0 | 7400 | Long et al. (2017) | Q | 287 |
| 2,3,3',4,6-pentabromodiphenyl ether C$_{12}$H$_5$Br$_5$O (PBDE-109) [446254-72-4] FXXXWTMLIQLDRP-UHFFFAOYSA-N | 1.2 | 7400 | Long et al. (2017) | Q | 287 |
| 2,3,3',4',6-pentabromodiphenyl ether C$_{12}$H$_5$Br$_5$O (PBDE-110) [446254-73-5] LESZGJVTZILBTK-UHFFFAOYSA-N | $9.4\times10^{-1}$ | 7400 | Long et al. (2017) | Q | 287 |



Table A7.2: Polybrominated diphenyl ethers (PBDEs) (...continued)

| Substance Formula (Trivial Name) [CAS Registry Number] InChIKey | $H_s^{cp}$ (at $T^\ominus$) $\left[\dfrac{\text{mol}}{\text{m}^3\,\text{Pa}}\right]$ | $\dfrac{\text{d}\ln H_s^{cp}}{\text{d}(1/T)}$ [K] | Reference | Type | Note |
|---|---|---|---|---|---|
| 2,3,3',5,5'-pentabromodiphenyl ether C$_{12}$H$_5$Br$_5$O (PBDE-111) [446254-74-6] PCHDCOXHJBWEPW-UHFFFAOYSA-N | 2.1 | 7400 | Long et al. (2017) | Q | 287 |
| 2,3,3',5,6-pentabromodiphenyl ether C$_{12}$H$_5$Br$_5$O (PBDE-112) [446254-75-7] MFBMNSFADPTAKZ-UHFFFAOYSA-N | 1.7 | 7400 | Long et al. (2017) | Q | 287 |
| 2,3,3',5',6-pentabromodiphenyl ether C$_{12}$H$_5$Br$_5$O (PBDE-113) [446254-76-8] OGZHLJXRGZFVLI-UHFFFAOYSA-N | 3.1 | 7400 | Long et al. (2017) | Q | 287 |
| 2,3,4,4',5-pentabromodiphenyl ether C$_{12}$H$_5$Br$_5$O (PBDE-114) [446254-77-9] SFNAUTSNWPPDSY-UHFFFAOYSA-N | $6.2\times10^{-1}$ $7.7\times10^{-1}$ $8.8\times10^{-1}$ $7.5\times10^{-1}$ | 4000 7400 | Lau et al. (2006) Lau et al. (2006) Charles and Destaillats (2005) Long et al. (2017) | M M M Q | 719 720 42 287 |
| 2,3,4,4',6-pentabromodiphenyl ether C$_{12}$H$_5$Br$_5$O (PBDE-115) [446254-78-0] BKTLDVXDOVSTEV-UHFFFAOYSA-N | $9.8\times10^{-1}$ | 7400 | Long et al. (2017) | Q | 287 |
| 2,3,4,5,6-pentabromodiphenyl ether C$_{12}$H$_5$Br$_5$O (PBDE-116) [189084-65-9] ACRQLFSHISNWRY-UHFFFAOYSA-N | $6.1\times10^{-1}$ | 7400 | Long et al. (2017) | Q | 287 |
| 2,3,4',5,6-pentabromodiphenyl ether C$_{12}$H$_5$Br$_5$O (PBDE-117) [446254-79-1] SOJBOGWFDBDWEG-UHFFFAOYSA-N | $8.6\times10^{-1}$ | 7400 | Long et al. (2017) | Q | 287 |



Table A7.2: Polybrominated diphenyl ethers (PBDEs) (...continued)

| Substance Formula (Trivial Name) [CAS Registry Number] InChIKey | $H_s^{cp}$ (at $T^\ominus$) $\left[\dfrac{\text{mol}}{\text{m}^3\,\text{Pa}}\right]$ | $\dfrac{\text{d}\ln H_s^{cp}}{\text{d}(1/T)}$ [K] | Reference | Type | Note |
|---|---|---|---|---|---|
| 2,3',4,4',5-pentabromodiphenyl ether C$_{12}$H$_5$Br$_5$O (PBDE-118) [446254-80-4] VTMFEPLDDHZBGI-UHFFFAOYSA-N | $9.2\times10^{-1}$ | 7400 | Long et al. (2017) | Q | 287 |
| 2,3',4,4',6-pentabromodiphenyl ether C$_{12}$H$_5$Br$_5$O (PBDE-119) [189084-66-0] KXEOYBYEJCRPGB-UHFFFAOYSA-N | 1.8 | 7400 | Long et al. (2017) | Q | 287 |
| 2,3',4,5,5'-pentabromodiphenyl ether C$_{12}$H$_5$Br$_5$O (PBDE-120) [417727-71-0] AKSBEUHDCRZJAN-UHFFFAOYSA-N | 1.7 | 7400 | Long et al. (2017) | Q | 287 |
| 2,3',4,5',6-pentabromodiphenyl ether C$_{12}$H$_5$Br$_5$O (PBDE-121) [446254-81-5] GVGNVZBJVFDAAO-UHFFFAOYSA-N | 5.9 | 7400 | Long et al. (2017) | Q | 287 |
| 2,3,3',4',5-pentabromodiphenyl ether C$_{12}$H$_5$Br$_5$O (PBDE-122) [446254-82-6] CDNHGSPFIUITTN-UHFFFAOYSA-N | $7.2\times10^{-1}$ | 7400 | Long et al. (2017) | Q | 287 |
| 2,3',4,4',5'-pentabromodiphenyl ether C$_{12}$H$_5$Br$_5$O (PBDE-123) [446254-83-7] SBKMUEQNZNDYFW-UHFFFAOYSA-N | 1.2 | 7400 | Long et al. (2017) | Q | 287 |
| 2,3',4',5,5'-pentabromodiphenyl ether C$_{12}$H$_5$Br$_5$O (PBDE-124) [446254-84-8] FGHJTAAHIFEHLT-UHFFFAOYSA-N | 1.1 | 7400 | Long et al. (2017) | Q | 287 |





Table A7.2: Polybrominated diphenyl ethers (PBDEs) (. . . continued)

| Substance Formula (Trivial Name) [CAS Registry Number] InChIKey | $H_s^{cp}$ (at $T^\ominus$) $\left[\dfrac{\text{mol}}{\text{m}^3\,\text{Pa}}\right]$ | $\dfrac{\text{d}\ln H_s^{cp}}{\text{d}(1/T)}$ [K] | Reference | Type | Note |
|---|---|---|---|---|---|
| 2,3',4',5',6-pentabromodiphenyl ether C$_{12}$H$_5$Br$_5$O (PBDE-125) [446254-85-9] SESXKFPOVUVGLR-UHFFFAOYSA-N | 1.3 | 7400 | Long et al. (2017) | Q | 287 |
| 3,3',4,4',5-pentabromodiphenyl ether C$_{12}$H$_5$Br$_5$O (PBDE-126) [366791-32-4] SJNIIWPIAVQNRK-UHFFFAOYSA-N | $5.6\times10^{-1}$ | 7400 | Long et al. (2017) | Q | 287 |
| 3,3',4,5,5'-pentabromodiphenyl ether C$_{12}$H$_5$Br$_5$O (PBDE-127) [446254-86-0] RATMRXKBPDCKCZ-UHFFFAOYSA-N | 1.1 | 7400 | Long et al. (2017) | Q | 287 |
| 2,2',3,3',4,4'-hexabromodiphenyl ether C$_{12}$H$_4$Br$_6$O (PBDE-128) [182677-28-7] WFLVELCLEGVBIH-UHFFFAOYSA-N | 2.1 | 7400 | Long et al. (2017) | Q | 287 |
| 2,2',3,3',4,5-hexabromodiphenyl ether C$_{12}$H$_4$Br$_6$O (PBDE-129) [446254-87-1] PRNCVYAUCSGSOE-UHFFFAOYSA-N | 1.9 | 7400 | Long et al. (2017) | Q | 287 |
| 2,2',3,3',4,5'-hexabromodiphenyl ether C$_{12}$H$_4$Br$_6$O (PBDE-130) [446254-88-2] YURCHLXPAGSJHU-UHFFFAOYSA-N | 3.7 | 7400 | Long et al. (2017) | Q | 287 |
| 2,2',3,3',4,6-hexabromodiphenyl ether C$_{12}$H$_4$Br$_6$O (PBDE-131) [446254-89-3] MGKVPJFIGGBCBA-UHFFFAOYSA-N | $1.0\times10^{1}$ | 7400 | Long et al. (2017) | Q | 287 |



Table A7.2: Polybrominated diphenyl ethers (PBDEs) (...continued)

| Substance<br>Formula<br>(Trivial Name)<br>[CAS Registry Number]<br>InChIKey | $H_s^{cp}$<br>(at $T^\ominus$)<br>$\left[\dfrac{\text{mol}}{\text{m}^3\,\text{Pa}}\right]$ | $\dfrac{\text{d}\ln H_s^{cp}}{\text{d}(1/T)}$<br><br>[K] | Reference | Type | Note |
|---|---|---|---|---|---|
| 2,2',3,3',4,6'-hexabromodiphenyl ether<br>$C_{12}H_4Br_6O$<br>(PBDE-132)<br>[446254-90-6]<br>FFEKBOKDYRZGRV-UHFFFAOYSA-N | 4.0 | 7400 | Long et al. (2017) | Q | 287 |
| 2,2',3,3',5,5'-hexabromodiphenyl ether<br>$C_{12}H_4Br_6O$<br>(PBDE-133)<br>[446254-91-7]<br>XTBFPFHQPGZZJX-UHFFFAOYSA-N | 6.3 | 7400 | Long et al. (2017) | Q | 287 |
| 2,2',3,3',5,6-hexabromodiphenyl ether<br>$C_{12}H_4Br_6O$<br>(PBDE-134)<br>[446254-92-8]<br>MIBDGPWSGDWIQR-UHFFFAOYSA-N | 4.8 | 7400 | Long et al. (2017) | Q | 287 |
| 2,2',3,3',5,6'-hexabromodiphenyl ether<br>$C_{12}H_4Br_6O$<br>(PBDE-135)<br>[446254-93-9]<br>AMGHASDTWACNCS-UHFFFAOYSA-N | 8.6 | 7400 | Long et al. (2017) | Q | 287 |
| 2,2',3,3',6,6'-hexabromodiphenyl ether<br>$C_{12}H_4Br_6O$<br>(PBDE-136)<br>[446254-94-0]<br>NTWGDSLWLUPCDW-UHFFFAOYSA-N | 7.8 | 7400 | Long et al. (2017) | Q | 287 |
| 2,2',3,4,4',5-hexabromodiphenyl ether<br>$C_{12}H_4Br_6O$<br>(PBDE-137)<br>[446254-95-1]<br>HSTYYNPYXZYIAG-UHFFFAOYSA-N | 3.0 | 7400 | Long et al. (2017) | Q | 287 |
| 2,2',3,4,4',5'-hexabromodiphenyl ether<br>$C_{12}H_4Br_6O$<br>(PBDE-138)<br>[182677-30-1]<br>IZFQCEZFGCMHOM-UHFFFAOYSA-N | 2.7 | 7400 | Long et al. (2017) | Q | 287 |



Table A7.2: Polybrominated diphenyl ethers (PBDEs) (. . . continued)

| Substance Formula (Trivial Name) [CAS Registry Number] InChIKey | $H_s^{cp}$ (at $T^{\ominus}$) $\left[\dfrac{\mathrm{mol}}{\mathrm{m^3\,Pa}}\right]$ | $\dfrac{\mathrm{d}\ln H_s^{cp}}{\mathrm{d}(1/T)}$ [K] | Reference | Type | Note |
|---|---|---|---|---|---|
| 2,2',3,4,4',6'-hexabromodiphenyl ether C$_{12}$H$_4$Br$_6$O (PBDE-140) [243982-83-4] FLRODCDHJZNIGA-UHFFFAOYSA-N | 7.7 | 7400 | Long et al. (2017) | Q | 287 |
| 2,2',3,4,5,5'-hexabromodiphenyl ether C$_{12}$H$_4$Br$_6$O (PBDE-141) [446254-97-3] XTXIYMGRRUJOIT-UHFFFAOYSA-N | 2.6 | 7400 | Long et al. (2017) | Q | 287 |
| 2,2',3,4,5,6-hexabromodiphenyl ether C$_{12}$H$_4$Br$_6$O (PBDE-142) [446254-98-4] LJDGJCNHVGGOFW-UHFFFAOYSA-N | 1.4 | 7400 | Long et al. (2017) | Q | 287 |
| 2,2',3,4,5,6'-hexabromodiphenyl ether C$_{12}$H$_4$Br$_6$O (PBDE-143) [446254-99-5] RQLZDUSZXOOBTM-UHFFFAOYSA-N | 5.1 | 7400 | Long et al. (2017) | Q | 287 |
| 2,2',3,4,5',6-hexabromodiphenyl ether C$_{12}$H$_4$Br$_6$O (PBDE-144) [446255-00-1] ZMSJCQOCTPYCQP-UHFFFAOYSA-N | 5.2 | 7400 | Long et al. (2017) | Q | 287 |
| 2,2',3,4,6,6'-hexabromodiphenyl ether C$_{12}$H$_4$Br$_6$O (PBDE-145) [446255-01-2] BTKLHMBWCRVCLC-UHFFFAOYSA-N | $1.0\times10^1$ | 7400 | Long et al. (2017) | Q | 287 |
| 2,2',3,4',5,5'-hexabromodiphenyl ether C$_{12}$H$_4$Br$_6$O (PBDE-146) [446255-02-3] HGXPYDNHBUCRTR-UHFFFAOYSA-N | 5.3 | 7400 | Long et al. (2017) | Q | 287 |





Table A7.2: Polybrominated diphenyl ethers (PBDEs) (...continued)

| Substance<br>Formula<br>(Trivial Name)<br>[CAS Registry Number]<br>InChIKey | $H_s^{cp}$<br>(at $T^\ominus$)<br><br>$\left[\dfrac{\text{mol}}{\text{m}^3\,\text{Pa}}\right]$ | $\dfrac{\text{d}\ln H_s^{cp}}{\text{d}(1/T)}$<br><br>[K] | Reference | Type | Note |
|---|---|---|---|---|---|
| 2,2',3,4',5,6-hexabromodiphenyl ether<br>$C_{12}H_4Br_6O$<br>(PBDE-147)<br>[116995-33-6]<br>OWBKWMDBTWHGHS-UHFFFAOYSA-N | 5.1 | 7400 | Long et al. (2017) | Q | 287 |
| 2,2',3,4',5,6'-hexabromodiphenyl ether<br>$C_{12}H_4Br_6O$<br>(PBDE-148)<br>[446255-03-4]<br>OJMHGSMSQZEBFH-UHFFFAOYSA-N | $1.2\times10^1$ | 7400 | Long et al. (2017) | Q | 287 |
| 2,2',3,4',5',6-hexabromodiphenyl ether<br>$C_{12}H_4Br_6O$<br>(PBDE-149)<br>[446255-04-5]<br>UJOUSZKYGGTPFQ-UHFFFAOYSA-N | 5.1 | 7400 | Long et al. (2017) | Q | 287 |
| 2,2',3,4',6,6'-hexabromodiphenyl ether<br>$C_{12}H_4Br_6O$<br>(PBDE-150)<br>[446255-05-6]<br>SQNOZOVDXXSLSG-UHFFFAOYSA-N | $1.7\times10^1$ | 7400 | Long et al. (2017) | Q | 287 |
| 2,2',3,5,5',6-hexabromodiphenyl ether<br>$C_{12}H_4Br_6O$<br>(PBDE-151)<br>[446255-06-7]<br>NGOQQUYCSISZMY-UHFFFAOYSA-N | 4.4 | 7400 | Long et al. (2017) | Q | 287 |
| 2,2',3,5,6,6'-hexabromodiphenyl ether<br>$C_{12}H_4Br_6O$<br>(PBDE-152)<br>[446255-07-8]<br>BYBJJARTBKUIJD-UHFFFAOYSA-N | 9.4 | 7400 | Long et al. (2017) | Q | 287 |
| 2,2',4,4',5,5'-hexabromodiphenyl ether<br>$C_{12}H_4Br_6O$<br>(PBDE-153)<br>[68631-49-2]<br>RZXIRSKYBISPGF-UHFFFAOYSA-N | 3.7 | 7200 | Long et al. (2017) | M | 287 |
|  | 3.5 | 7800 | Cetin and Odabasi (2005) | M |  |
|  | 6.1 |  | Kuramochi et al. (2014) | V |  |
|  | $1.5\times10^1$ |  | Tittlemier et al. (2002) | V |  |
|  | 2.9 |  | Wania and Dugani (2003) | R |  |
|  | 3.9 | 7400 | Long et al. (2017) | Q | 287 |
|  | $8.4\times10^{-1}$ |  | Hilal et al. (2008) | Q |  |



Table A7.2: Polybrominated diphenyl ethers (PBDEs) (... continued)

| Substance Formula (Trivial Name) [CAS Registry Number] InChIKey | $H_s^{cp}$ (at $T^\ominus$) $\left[\dfrac{\text{mol}}{\text{m}^3\,\text{Pa}}\right]$ | $\dfrac{\text{d}\ln H_s^{cp}}{\text{d}(1/T)}$ [K] | Reference | Type | Note |
|---|---|---|---|---|---|
| 2,2',4,4',5,6'-hexabromodiphenyl ether | 7.1 | 6800 | Long et al. (2017) | M | 287 |
| $C_{12}H_4Br_6O$ | 7.3 | 6800 | Cetin and Odabasi (2005) | M | |
| (PBDE-154) | 4.2 | | Tittlemier et al. (2002) | V | |
| [207122-15-4] | $1.1 \times 10^1$ | 7400 | Long et al. (2017) | Q | 287 |
| VHNPZYZQKWIWOD-UHFFFAOYSA-N | $7.2 \times 10^{-1}$ | | Hilal et al. (2008) | Q | |
| 2,2',4,4',6,6'-hexabromodiphenyl ether $C_{12}H_4Br_6O$ (PBDE-155) [35854-94-5] HRSCBOSGEKXXSI-UHFFFAOYSA-N | $2.5 \times 10^1$ | 7400 | Long et al. (2017) | Q | 287 |
| 2,3,3',4,4',5-hexabromodiphenyl ether $C_{12}H_4Br_6O$ (PBDE-156) [405237-85-6] JSDPCMJWYRDQEV-UHFFFAOYSA-N | 1.5 | 7400 | Long et al. (2017) | Q | 287 |
| 2,3,3',4,4',5'-hexabromodiphenyl ether $C_{12}H_4Br_6O$ (PBDE-157) [446255-08-9] JUOAMVUIJQJZSZ-UHFFFAOYSA-N | 1.9 | 7400 | Long et al. (2017) | Q | 287 |
| 2,3,3',4,4',6-hexabromodiphenyl ether $C_{12}H_4Br_6O$ (PBDE-158) [446255-09-0] KRYHHTVQOOJNHQ-UHFFFAOYSA-N | 2.7 | 7400 | Long et al. (2017) | Q | 287 |
| 2,3,3',4,5,5'-hexabromodiphenyl ether $C_{12}H_4Br_6O$ (PBDE-159) [446255-10-3] IDYFFNCFLRCOPZ-UHFFFAOYSA-N | 3.0 | 7400 | Long et al. (2017) | Q | 287 |
| 2,3,3',4,5,6-hexabromodiphenyl ether $C_{12}H_4Br_6O$ (PBDE-160) [446255-11-4] OCVOYHGOXIIONK-UHFFFAOYSA-N | 2.8 | 7400 | Long et al. (2017) | Q | 287 |





Table A7.2: Polybrominated diphenyl ethers (PBDEs) (... continued)

| Substance<br>Formula<br>(Trivial Name)<br>[CAS Registry Number]<br>InChIKey | $H_s^{cp}$<br>(at $T^\ominus$)<br>$\left[\dfrac{\text{mol}}{\text{m}^3\,\text{Pa}}\right]$ | $\dfrac{\text{d}\ln H_s^{cp}}{\text{d}(1/T)}$<br><br>[K] | Reference | Type | Note |
|---|---|---|---|---|---|
| 2,3,3',4,5',6-hexabromodiphenyl ether<br>$C_{12}H_4Br_6O$<br>(PBDE-161)<br>[446255-12-5]<br>WEYWRBBPPKSRGU-UHFFFAOYSA-N | 9.0 | 7400 | Long et al. (2017) | Q | 287 |
| 2,3,3',4',5,5'-hexabromodiphenyl ether<br>$C_{12}H_4Br_6O$<br>(PBDE-162)<br>[446255-13-6]<br>UKPNCLHMNJCGCJ-UHFFFAOYSA-N | 2.7 | 7400 | Long et al. (2017) | Q | 287 |
| 2,3,3',4',5,6-hexabromodiphenyl ether<br>$C_{12}H_4Br_6O$<br>(PBDE-163)<br>[446255-14-7]<br>NUEAHMLXQFHEJN-UHFFFAOYSA-N | 2.3 | 7400 | Long et al. (2017) | Q | 287 |
| 2,3,3',4',5',6-hexabromodiphenyl ether<br>$C_{12}H_4Br_6O$<br>(PBDE-164)<br>[446255-15-8]<br>UJVYVXIHTJOJBZ-UHFFFAOYSA-N | 3.1 | 7400 | Long et al. (2017) | Q | 287 |
| 2,3,3',5,5',6-hexabromodiphenyl ether<br>$C_{12}H_4Br_6O$<br>(PBDE-165)<br>[446255-16-9]<br>KXERERDGMTWBGZ-UHFFFAOYSA-N | 7.9 | 7400 | Long et al. (2017) | Q | 287 |
| 2,3,4,4',5,6-hexabromodiphenyl ether<br>$C_{12}H_4Br_6O$<br>(PBDE-166)<br>[189084-58-0]<br>KVYODBMKQYVNEK-UHFFFAOYSA-N | 1.2 | 7400 | Long et al. (2017) | Q | 287 |
| 2,3',4,4',5,5'-hexabromodiphenyl ether<br>$C_{12}H_4Br_6O$<br>(PBDE-167)<br>[446255-17-0]<br>NMUPLZRHSXJCJQ-UHFFFAOYSA-N | 3.1 | 7400 | Long et al. (2017) | Q | 287 |



Table A7.2: Polybrominated diphenyl ethers (PBDEs) (. . . continued)

| Substance Formula (Trivial Name) [CAS Registry Number] InChIKey | $H_s^{cp}$ (at $T^\ominus$) $\left[\dfrac{\mathrm{mol}}{\mathrm{m^3\,Pa}}\right]$ | $\dfrac{\mathrm{d}\ln H_s^{cp}}{\mathrm{d}(1/T)}$ [K] | Reference | Type | Note |
|---|---|---|---|---|---|
| 2,3',4,4',5',6-hexabromodiphenyl ether $C_{12}H_4Br_6O$ (PBDE-168) [53551-87-4] HWZAPXGFMVEGPW-UHFFFAOYSA-N | 5.0 | 7400 | Long et al. (2017) | Q | 287 |
| 3,3',4,4',5,5'-hexabromodiphenyl ether $C_{12}H_4Br_6O$ (PBDE-169) [446255-18-1] JKFBMDHBJYKFKL-UHFFFAOYSA-N | 1.2 | 7400 | Long et al. (2017) | Q | 287 |
| 2,2',3,3',4,4',5-heptabromodiphenyl ether $C_{12}H_3Br_7O$ (PBDE-170) [327185-13-7] DLPNCMQTNWLTHD-UHFFFAOYSA-N | 4.9 | 7400 | Long et al. (2017) | Q | 287 |
| 2,2',3,3',4,4',6-heptabromodiphenyl ether $C_{12}H_3Br_7O$ (PBDE-171) [446255-19-2] FRMMMROUUPQUMZ-UHFFFAOYSA-N | $1.2\times10^1$ | 7400 | Long et al. (2017) | Q | 287 |
| 2,2',3,3',4,5,5'-heptabromodiphenyl ether $C_{12}H_3Br_7O$ (PBDE-172) [407606-59-1] DSRRSKFMOJQETR-UHFFFAOYSA-N | $1.6\times10^1$ | 7400 | Long et al. (2017) | Q | 287 |
| 2,2',3,3',4,5,6-heptabromodiphenyl ether $C_{12}H_3Br_7O$ (PBDE-173) [446255-20-5] NLBLNZDNOSSGPW-UHFFFAOYSA-N | 6.7 | 7400 | Long et al. (2017) | Q | 287 |



Table A7.2: Polybrominated diphenyl ethers (PBDEs) (... continued)

| Substance<br>Formula<br>(Trivial Name)<br>[CAS Registry Number]<br>InChIKey | $H_s^{cp}$<br>(at $T^\ominus$)<br>$\left[\dfrac{\text{mol}}{\text{m}^3\,\text{Pa}}\right]$ | $\dfrac{\text{d}\ln H_s^{cp}}{\text{d}(1/T)}$<br><br>[K] | Reference | Type | Note |
|---|---|---|---|---|---|
| 2,2',3,3',4,5,6'-<br>heptabromodiphenyl<br>ether<br>$C_{12}H_3Br_7O$<br>(PBDE-174)<br>[446255-21-6]<br>VUUWOHUOYUGBEO-UHFFFAOYSA-N | $1.2 \times 10^1$ | 7400 | Long et al. (2017) | Q | 287 |
| 2,2',3,3',4,5',6-<br>heptabromodiphenyl<br>ether<br>$C_{12}H_3Br_7O$<br>(PBDE-175)<br>[6255-22-7]<br>YATZWTXATDYQCK-UHFFFAOYSA-N | $1.7 \times 10^1$ | 7400 | Long et al. (2017) | Q | 287 |
| 2,2',3,3',4,6,6'-<br>heptabromodiphenyl<br>ether<br>$C_{12}H_3Br_7O$<br>(PBDE-176)<br>[407606-61-5]<br>SWUALKCOTZOSMY-UHFFFAOYSA-N | $2.5 \times 10^1$ | 7400 | Long et al. (2017) | Q | 287 |
| 2,2',3,3',4,5',6'-<br>heptabromodiphenyl<br>ether<br>$C_{12}H_3Br_7O$<br>(PBDE-177)<br>[446255-23-8]<br>ZHUHLPXIJIBQBJ-UHFFFAOYSA-N | 9.9 | 7400 | Long et al. (2017) | Q | 287 |
| 2,2',3,3',5,5',6-<br>heptabromodiphenyl<br>ether<br>$C_{12}H_3Br_7O$<br>(PBDE-178)<br>[446255-24-9]<br>UWUVZUPEEORCRG-UHFFFAOYSA-N | $1.3 \times 10^1$ | 7400 | Long et al. (2017) | Q | 287 |
| 2,2',3,3',5,6,6'-<br>heptabromodiphenyl<br>ether<br>$C_{12}H_3Br_7O$<br>(PBDE-179)<br>[446255-25-0]<br>COVXWWKOLMNRQE-UHFFFAOYSA-N | $2.2 \times 10^1$ | 7400 | Long et al. (2017) | Q | 287 |





Table A7.2: Polybrominated diphenyl ethers (PBDEs) (...continued)

| Substance Formula (Trivial Name) [CAS Registry Number] InChIKey | $H_s^{cp}$ (at $T^\ominus$) $\left[\dfrac{\text{mol}}{\text{m}^3\,\text{Pa}}\right]$ | $\dfrac{\text{d}\ln H_s^{cp}}{\text{d}(1/T)}$ [K] | Reference | Type | Note |
|---|---|---|---|---|---|
| 2,2',3,4,4',5,5'-heptabromodiphenyl ether $C_{12}H_3Br_7O$ (PBDE-180) [446255-26-1] STMBXVOJNOJRPZ-UHFFFAOYSA-N | 7.3 | 7400 | Long et al. (2017) | Q | 287 |
| 2,2',3,4,4',5,6-heptabromodiphenyl ether $C_{12}H_3Br_7O$ (PBDE-181) [189084-67-1] GVNRIAPLVGNZPL-UHFFFAOYSA-N | 8.9 | 7400 | Long et al. (2017) | Q | 287 |
| 2,2',3,4,4',5,6'-heptabromodiphenyl ether $C_{12}H_3Br_7O$ (PBDE-182) [442690-45-1] ZYHDTADADSNMLV-UHFFFAOYSA-N | $1.9\times10^1$ | 7400 | Long et al. (2017) | Q | 287 |
| 2,2',3,4,4',5',6-heptabromodiphenyl ether $C_{12}H_3Br_7O$ (PBDE-183) [207122-16-5] ILPSCQCLBHQUEM-UHFFFAOYSA-N | $1.4\times10^2$ $1.3\times10^1$ | 7400 | Tittlemier et al. (2002) Long et al. (2017) | V Q | 287 |
| 2,2',3,4,4',6,6'-heptabromodiphenyl ether $C_{12}H_3Br_7O$ (PBDE-184) [117948-63-7] JHDCZVAQPRXHEL-UHFFFAOYSA-N | $3.7\times10^1$ | 7400 | Long et al. (2017) | Q | 287 |
| 2,2',3,4,5,5',6-heptabromodiphenyl ether $C_{12}H_3Br_7O$ (PBDE-185) [405237-86-7] YRNMIFAQDSUFTR-UHFFFAOYSA-N | 5.1 | 7400 | Long et al. (2017) | Q | 287 |





Table A7.2: Polybrominated diphenyl ethers (PBDEs) (...continued)

| Substance Formula (Trivial Name) [CAS Registry Number] InChIKey | $H_s^{cp}$ (at $T^\ominus$) $\left[\dfrac{\text{mol}}{\text{m}^3\,\text{Pa}}\right]$ | $\dfrac{\text{d}\ln H_s^{cp}}{\text{d}(1/T)}$ [K] | Reference | Type | Note |
|---|---|---|---|---|---|
| 2,2',3,4,5,6,6'-heptabromodiphenyl ether C$_{12}$H$_3$Br$_7$O (PBDE-186) [446255-27-2] WUFQDCMRKKDNSF-UHFFFAOYSA-N | $1.4\times10^1$ | 7400 | Long et al. (2017) | Q | 287 |
| 2,2',3,4',5,5',6-heptabromodiphenyl ether C$_{12}$H$_3$Br$_7$O (PBDE-187) [446255-28-3] RFZPXOBFDARWHV-UHFFFAOYSA-N | 9.4 | 7400 | Long et al. (2017) | Q | 287 |
| 2,2',3,4',5,6,6'-heptabromodiphenyl ether C$_{12}$H$_3$Br$_7$O (PBDE-188) [116995-32-5] YGYDHFDPVGAMTL-UHFFFAOYSA-N | $3.5\times10^1$ | 7400 | Long et al. (2017) | Q | 287 |
| 2,3,3',4,4',5,5'-heptabromodiphenyl ether C$_{12}$H$_3$Br$_7$O (PBDE-189) [259087-35-9] CQVLRTUESBMMJW-UHFFFAOYSA-N | 4.4 | 7400 | Long et al. (2017) | Q | 287 |
| 2,3,3',4,4',5,6-heptabromodiphenyl ether C$_{12}$H$_3$Br$_7$O (PBDE-190) [189084-68-2] OUEYHQIMJGHOQN-UHFFFAOYSA-N | 4.0 | 7400 | Long et al. (2017) | Q | 287 |
| 2,3,3',4,4',5',6-heptabromodiphenyl ether C$_{12}$H$_3$Br$_7$O (PBDE-191) [446255-30-7] BNBFKFHSIPERIM-UHFFFAOYSA-N | 8.4 | 7400 | Long et al. (2017) | Q | 287 |





Table A7.2: Polybrominated diphenyl ethers (PBDEs) (...continued)

| Substance Formula (Trivial Name) [CAS Registry Number] InChIKey | $H_s^{cp}$ (at $T^\ominus$) $\left[\dfrac{\text{mol}}{\text{m}^3\,\text{Pa}}\right]$ | $\dfrac{\text{d}\ln H_s^{cp}}{\text{d}(1/T)}$ [K] | Reference | Type | Note |
|---|---|---|---|---|---|
| 2,3,3',4,5,5',6-heptabromodiphenyl ether C$_{12}$H$_3$Br$_7$O (PBDE-192) [407578-53-4] ABLZOLAUBUSUHT-UHFFFAOYSA-N | $1.3\times10^1$ | 7400 | Long et al. (2017) | Q | 287 |
| 2,3,3',4',5,5',6-heptabromodiphenyl ether C$_{12}$H$_3$Br$_7$O (PBDE-193) [446255-34-1] AUFJSWANTKXCFZ-UHFFFAOYSA-N | 7.3 | 7400 | Long et al. (2017) | Q | 287 |
| 2,2',3,3',4,4',5,5'-octabromodiphenyl ether C$_{12}$H$_2$Br$_8$O (PBDE-194) [32536-52-0] ORYGKUIDIMIRNN-UHFFFAOYSA-N | $1.3\times10^1$ | 7400 | Long et al. (2017) | Q | 287 |
| 2,2',3,3',4,4',5,6-octabromodiphenyl ether C$_{12}$H$_2$Br$_8$O (PBDE-195) [446255-38-5] GPQLSLKPHQEEOP-UHFFFAOYSA-N | $1.6\times10^1$ | 7400 | Long et al. (2017) | Q | 287 |
| 2,2',3,3',4,4',5,6'-octabromodiphenyl ether C$_{12}$H$_2$Br$_8$O (PBDE-196) [446255-39-6] IEWFKOVTVJNWFF-UHFFFAOYSA-N | $3.7\times10^1$ | 7400 | Long et al. (2017) | Q | 287 |
| 2,2',3,3',4,4',6,6'-octabromodiphenyl ether C$_{12}$H$_2$Br$_8$O (PBDE-197) [117964-21-3] AAFUUKPTSPVXJH-UHFFFAOYSA-N | $7.1\times10^1$ | 7400 | Long et al. (2017) | Q | 287 |



Table A7.2: Polybrominated diphenyl ethers (PBDEs) (...continued)

| Substance Formula (Trivial Name) [CAS Registry Number] InChIKey | $H_s^{cp}$ (at $T^{\ominus}$) $\left[\dfrac{\text{mol}}{\text{m}^3\,\text{Pa}}\right]$ | $\dfrac{\text{d}\ln H_s^{cp}}{\text{d}(1/T)}$ [K] | Reference | Type | Note |
|---|---|---|---|---|---|
| 2,2',3,3',4,5,5',6-octabromodiphenyl ether $C_{12}H_2Br_8O$ (PBDE-198) [446255-42-1] IBKRHVDFFHQOSC-UHFFFAOYSA-N | $1.8\times10^1$ | 7400 | Long et al. (2017) | Q | 287 |
| 2,2',3,3',4,5,5',6'-octabromodiphenyl ether $C_{12}H_2Br_8O$ (PBDE-199) [446255-43-2] JNSLJYRXDGBNBE-UHFFFAOYSA-N | $3.3\times10^1$ | 7400 | Long et al. (2017) | Q | 287 |
| 2,2',3,3',4,5,6,6'-octabromodiphenyl ether $C_{12}H_2Br_8O$ (PBDE-200) [446255-46-5] JWMXGEPFVCRXQR-UHFFFAOYSA-N | $3.7\times10^1$ | 7400 | Long et al. (2017) | Q | 287 |
| 2,2',3,3',4,5',6,6'-octabromodiphenyl ether $C_{12}H_2Br_8O$ (PBDE-201) [446255-50-1] HQWFMMKREWXIGN-UHFFFAOYSA-N | $6.8\times10^1$ | 7400 | Long et al. (2017) | Q | 287 |
| 2,2',3,3',5,5',6,6'-octabromodiphenyl ether $C_{12}H_2Br_8O$ (PBDE-202) [67797-09-5] AHNZLQAZTWRRDW-UHFFFAOYSA-N | $5.7\times10^1$ | 7400 | Long et al. (2017) | Q | 287 |
| 2,2',3,4,4',5,5',6-octabromodiphenyl ether $C_{12}H_2Br_8O$ (PBDE-203) [337513-72-1] RTUZOQFRIPIWPS-UHFFFAOYSA-N | $2.0\times10^1$ | 7400 | Long et al. (2017) | Q | 287 |



Table A7.2: Polybrominated diphenyl ethers (PBDEs) (...continued)

| Substance<br>Formula<br>(Trivial Name)<br>[CAS Registry Number]<br>InChIKey | $H_s^{cp}$<br>(at $T^\ominus$)<br>$\left[\dfrac{\mathrm{mol}}{\mathrm{m^3\,Pa}}\right]$ | $\dfrac{\mathrm{d}\ln H_s^{cp}}{\mathrm{d}(1/T)}$<br><br>[K] | Reference | Type | Note |
|---|---|---|---|---|---|
| 2,2',3,4,4',5,6,6'-<br>octabromodiphenyl<br>ether<br>$C_{12}H_2Br_8O$<br>(PBDE-204)<br>[446255-54-5]<br>YZABCBOJTHQTSX-UHFFFAOYSA-N | $6.2 \times 10^1$ | 7400 | Long et al. (2017) | Q | 287 |
| 2,3,3',4,4',5,5',6-<br>octabromodiphenyl<br>ether<br>$C_{12}H_2Br_8O$<br>(PBDE-205)<br>[446255-56-7]<br>CVMKCYDBEYHNBM-UHFFFAOYSA-N | $1.8 \times 10^1$ | 7400 | Long et al. (2017) | Q | 287 |
| 2,2',3,3',4,4',5,5',6-<br>nonabromodiphenyl<br>ether<br>$C_{12}HBr_9O$<br>(PBDE-206)<br>[63387-28-0]<br>CYRHBNRLQMLULE-UHFFFAOYSA-N | $4.7 \times 10^1$ | 7400 | Long et al. (2017) | Q | 287 |
| 2,2',3,3',4,4',5,6,6'-<br>nonabromodiphenyl<br>ether<br>$C_{12}HBr_9O$<br>(PBDE-207)<br>[437701-79-6]<br>IEEVDIAVLGLVOW-UHFFFAOYSA-N | $1.2 \times 10^2$ | 7400 | Long et al. (2017) | Q | 287 |
| 2,2',3,3',4,5,5',6,6'-<br>nonabromodiphenyl<br>ether<br>$C_{12}HBr_9O$<br>(PBDE-208)<br>[437701-78-5]<br>ASGZXYIDLFWXID-UHFFFAOYSA-N | $1.1 \times 10^2$ | 7400 | Long et al. (2017) | Q | 287 |
| decabromodiphenyl ether<br>$C_{12}Br_{10}O$<br>(PBDE-209)<br>[1163-19-5]<br>WHHGLZMJPXIBIX-UHFFFAOYSA-N | $1.8 \times 10^1$<br>$1.8 \times 10^1$<br>$2.7 \times 10^1$<br>$8.2 \times 10^2$<br>$8.2 \times 10^2$<br>$4.1 \times 10^2$<br>$1.3 \times 10^3$<br>$6.7 \times 10^2$ | 7600<br>7900<br>7400 | Long et al. (2017)<br>Cetin and Odabasi (2005)<br>Long et al. (2017)<br>HSDB (2015)<br>Zhang et al. (2010)<br>Zhang et al. (2010)<br>Zhang et al. (2010)<br>Zhang et al. (2010) | M<br>M<br>Q<br>Q<br>Q<br>Q<br>Q<br>Q | 287<br><br>287<br>99<br>287, 288<br>287, 289<br>287, 290<br>287, 291 |



Table A7.2: Polybrominated diphenyl ethers (PBDEs) (. . . continued)

| Substance Formula (Trivial Name) [CAS Registry Number] InChIKey | $H_s^{cp}$ (at $T^{\ominus}$) $\left[\dfrac{\mathrm{mol}}{\mathrm{m^3\,Pa}}\right]$ | $\dfrac{\mathrm{d}\ln H_s^{cp}}{\mathrm{d}(1/T)}$ [K] | Reference | Type | Note |
|---|---|---|---|---|---|
| dibromoacetonitrile $C_2HBr_2N$ [3252-43-5] NDSBDLSWTGLNQA-UHFFFAOYSA-N | $2.4\times10^1$ | | HSDB (2015) | Q | 99 |
| bromoacetonitrile $C_2H_2BrN$ [590-17-0] REXUYBKPWIPONM-UHFFFAOYSA-N | 2.8 | | HSDB (2015) | Q | 447 |
| 2-bromopyridine $C_5H_4BrN$ [109-04-6] IMRWILPUOVGIMU-UHFFFAOYSA-N | 1.3 $1.8\times10^{-1}$ | | Duchowicz et al. (2020) Duchowicz et al. (2020) | V Q | 186 |
| 3-bromopyridine $C_5H_4BrN$ [626-55-1] NYPYPOZNGOXYSU-UHFFFAOYSA-N | $8.3\times10^{-1}$ | | Ebert et al. (2023) | ? | 316 |
| 1,2-dibromo-2,4-dicyanobutane $C_6H_6Br_2N_2$ [35691-65-7] DHVLDKHFGIVEIP-UHFFFAOYSA-N | $1.2\times10^3$ | | HSDB (2015) | V | |
| 4-bromobenzenamine $C_6H_6BrN$ [106-40-1] WDFQBORIUYODSI-UHFFFAOYSA-N | $1.1\times10^1$ | | HSDB (2015) | Q | 447 |
| 2,4,6-tribromobenzenamine $C_6H_4Br_3N$ [147-82-0] GVPODVKBTHCGFU-UHFFFAOYSA-N | $8.2\times10^1$ 2.6 $6.0\times10^{-1}$ $1.2\times10^2$ | | Zhang et al. (2010) Zhang et al. (2010) Zhang et al. (2010) Zhang et al. (2010) | Q Q Q Q | 287, 288 287, 289 287, 290 287, 291 |
| N,N'-dimethyl-3,3',4,4',5,5'- hexabromo-2,2'-bipyrrole $C_{10}H_6Br_6N_2$ (DBP-Br6) [253798-63-9] BUKWTHPBLJVYMZ-UHFFFAOYSA-N | $5.0\times10^2$ $5.1\times10^1$ | | Tittlemier et al. (2004) Hilal et al. (2008) | V Q | |
| 1,1'-ethylene 2,2'-dipyridylium dibromide $C_{12}H_{12}N_2Br$ (diquat dibromide) [85-00-7] JXEXEPZXXFNEHA-UHFFFAOYSA-M | $7.0\times10^7$ | | HSDB (2015) | Q | 99 |



Table A7.2: Polybrominated diphenyl ethers (PBDEs) (...continued)

| Substance<br>Formula<br>(Trivial Name)<br>[CAS Registry Number]<br>InChIKey | $H_s^{cp}$<br>(at $T^\ominus$)<br>$\left[\dfrac{\mathrm{mol}}{\mathrm{m^3\,Pa}}\right]$ | $\dfrac{\mathrm{d}\ln H_s^{cp}}{\mathrm{d}(1/T)}$<br><br>[K] | Reference | Type | Note |
|---|---|---|---|---|---|
| tralomethrin<br>$C_{22}H_{19}NO_3Br_4$<br>[66841-25-6]<br>YWSCPYYRJXKUDB-KAKFPZCNSA-N | $2.5\times10^4$<br>$2.5\times10^4$<br>$1.2\times10^3$ | | Duchowicz et al. (2020)<br>HSDB (2015)<br>Duchowicz et al. (2020) | V<br>V<br>Q | 186 |
| bromomethyl peroxynitrate<br>$CH_2BrO_2NO_2$<br>JTAFUKGUWIYDNI-UHFFFAOYSA-N | $3.5\times10^{-1}$ | | Krysztofiak et al. (2012) | Q | |
| dibromomethyl peroxynitrate<br>$CHBr_2O_2NO_2$<br>CRBGJNSRTFAVGH-UHFFFAOYSA-N | 3.0 | | Krysztofiak et al. (2012) | Q | |
| tribromomethyl peroxynitrate<br>$CBr_3O_2NO_2$<br>BVMOLEXBJGZUAI-UHFFFAOYSA-N | 4.0 | | Krysztofiak et al. (2012) | Q | |
| MCM:BRETPAN<br>$C_2H_2NO_5Br$<br>OOFSXJVGALRPLK-UHFFFAOYSA-N | $2.0\times10^1$<br>$8.7\times10^1$<br>$3.3\times10^{-3}$ | | Wang et al. (2017)<br>Wang et al. (2017)<br>Wang et al. (2017) | Q<br>Q<br>Q | 80, 238<br>80, 239<br>80, 240 |
| 2,2-dibromo-2-cyanoacetamide<br>$C_3H_2Br_2N_2O$<br>(2,2-dibromo-3-<br>nitrilopropionamide)<br>[10222-01-2]<br>UUIVKBHZENILKB-UHFFFAOYSA-N | $5.2\times10^2$<br>$5.2\times10^2$<br>$8.5\times10^3$ | | Duchowicz et al. (2020)<br>HSDB (2015)<br>Duchowicz et al. (2020) | V<br>V<br>Q | 186 |
| bronopol<br>$C_3H_6BrNO_4$<br>[52-51-7]<br>LVDKZNITIUWNER-UHFFFAOYSA-N | $7.6\times10^5$ | | HSDB (2015) | V | |
| 2,6-dibromo-4-nitroaniline<br>$C_6H_4Br_2N_2O_2$<br>[827-94-1]<br>YMZIFDLWYUSZCC-UHFFFAOYSA-N | $8.2\times10^3$<br>$1.7\times10^2$<br>$1.9\times10^3$<br>$5.7\times10^3$ | | Zhang et al. (2010)<br>Zhang et al. (2010)<br>Zhang et al. (2010)<br>Zhang et al. (2010) | Q<br>Q<br>Q<br>Q | 287, 288<br>287, 289<br>287, 290<br>287, 291 |
| 2-bromo-4,6-dinitroaniline<br>$C_6H_4BrN_3O_4$<br>[1817-73-8]<br>KWMDHCLJYMVBNS-UHFFFAOYSA-N | $3.9\times10^4$<br>$2.7\times10^2$<br>$1.8\times10^3$<br>$5.7\times10^4$ | | Zhang et al. (2010)<br>Zhang et al. (2010)<br>Zhang et al. (2010)<br>Zhang et al. (2010) | Q<br>Q<br>Q<br>Q | 287, 288<br>287, 289<br>287, 290<br>287, 291 |
| 3-bromonitrobenzene<br>$C_6H_4BrNO_2$<br>($m$-bromonitrobenzene)<br>[585-79-5]<br>FWIROFMBWVMWLB-UHFFFAOYSA-N | 5.3<br>5.4<br>1.8 | | Duchowicz et al. (2020)<br>Schüürmann (2000)<br>Duchowicz et al. (2020) | V<br>V<br>Q | 186 |



Table A7.2: Polybrominated diphenyl ethers (PBDEs) (... continued)

| Substance Formula (Trivial Name) [CAS Registry Number] InChIKey | $H_s^{cp}$ (at $T^\ominus$) $\left[\dfrac{\text{mol}}{\text{m}^3\,\text{Pa}}\right]$ | $\dfrac{\text{d}\ln H_s^{cp}}{\text{d}(1/T)}$ [K] | Reference | Type | Note |
|---|---|---|---|---|---|
| 4-bromonitrobenzene $C_6H_4BrNO_2$ (*p*-bromonitrobenzene) [586-78-7] ZDFBKZUDCQQKAC-UHFFFAOYSA-N | $2.2\times10^{-1}$ | | Li et al. (2014) | Q | 241 |
| 3,5-dibromo-4-hydroxy-benzonitrile $C_7H_3Br_2NO$ (bromoxynil) [1689-84-5] UPMXNNIRAGDFEH-UHFFFAOYSA-N | $7.4\times10^2$ $1.0\times10^6$ $2.2\times10^3$ $1.1\times10^6$ | | Mackay et al. (2006d) Maniere et al. (2011) Maniere et al. (2011) Maniere et al. (2011) | V ? ? ? | 241, 573, 165 241, 577, 165 241, 493, 165 |
| 2,6-dibromo-3-methyl-4-nitroanisole $C_8H_7Br_2NO_3$ [62265-99-0] RBAJFFLBHVZCDY-UHFFFAOYSA-N | $4.5\times10^1$ $3.5\times10^1$ 4.7 9.7 | | Zhang et al. (2010) Zhang et al. (2010) Zhang et al. (2010) Zhang et al. (2010) | Q Q Q Q | 287, 288 287, 289 287, 290 287, 291 |
| bromacil $C_9H_{13}BrN_2O_2$ [314-40-9] CTSLUCNDVMMDHG-UHFFFAOYSA-N | $7.6\times10^4$ $7.8\times10^4$ $5.3\times10^2$ 5.2 $2.1\times10^1$ | | HSDB (2015) Mackay et al. (2006d) Suntio et al. (1988) Barcelo and Hennion (1997) Goodarzi et al. (2010) | V V V X Q | 12 567 568 |
| N'-(4-bromophenyl)-N-methoxy-N-methylurea $C_9H_{11}BrN_2O$ (metobromuron) [3060-89-7] WLFDQEVORAMCIM-UHFFFAOYSA-N | $3.2\times10^3$ $3.2\times10^3$ $3.2\times10^1$ 8.5 $8.8\times10^3$ | | HSDB (2015) Mackay et al. (2006d) Barcelo and Hennion (1997) Goodarzi et al. (2010) Maniere et al. (2011) | V V X Q ? | 567 568 12, 165 |
| bromuron $C_9H_{11}BrN_2O$ [3408-97-7] GSNZNZUNAJCHDO-UHFFFAOYSA-N | $2.0\times10^4$ | | MacBean (2012a) | ? | |
| bromoxynil butyrate $C_{11}H_9Br_2NO_2$ [3861-41-4] PGMZYNZXIYOOHJ-UHFFFAOYSA-N | $5.6\times10^1$ | | Maniere et al. (2011) | ? | 12, 165 |
| tris(2,3-dibromopropyl)isocyanurate $C_{12}H_{15}Br_6N_3O_3$ [52434-90-9] NZUPFZNVGSWLQC-UHFFFAOYSA-N | $8.2\times10^{12}$ $6.5\times10^7$ $2.7\times10^8$ $1.2\times10^{10}$ | | Zhang et al. (2010) Zhang et al. (2010) Zhang et al. (2010) Zhang et al. (2010) | Q Q Q Q | 287, 288 287, 289 287, 290 287, 291 |



Table A7.2: Polybrominated diphenyl ethers (PBDEs) (. . . continued)

| Substance Formula (Trivial Name) [CAS Registry Number] InChIKey | $H_s^{cp}$ (at $T^{\ominus}$) $\left[\dfrac{\mathrm{mol}}{\mathrm{m}^3\,\mathrm{Pa}}\right]$ | $\dfrac{\mathrm{d}\ln H_s^{cp}}{\mathrm{d}(1/T)}$ [K] | Reference | Type | Note |
|---|---|---|---|---|---|
| bromofenoxim $C_{13}H_7Br_2N_3O_6$ [13181-17-4] VTQWKUZUPOTXEH-UHFFFAOYSA-N | $3.1\times10^{-3}$ $8.1\times10^{-4}$ $1.3\times10^{5}$ | | Barcelo and Hennion (1997) Goodarzi et al. (2010) MacBean (2012a) | X Q ? | 567 568, 569 12 |
| tribromsalan $C_{13}H_8Br_3NO_2$ [87-10-5] KVSKGMLNBAPGKH-UHFFFAOYSA-N | $9.7\times10^{5}$ $1.2\times10^{6}$ $1.6\times10^{6}$ $1.2\times10^{6}$ | | Zhang et al. (2010) Zhang et al. (2010) Zhang et al. (2010) Zhang et al. (2010) | Q Q Q Q | 287, 288 287, 289 287, 290 287, 291 |
| bromoxynil heptanoate $C_{14}H_{15}Br_2NO_2$ [56634-95-8] BHZWBQPHPLFZSV-UHFFFAOYSA-N | 5.3 | | Maniere et al. (2011) | ? | 12, 165 |
| 1-amino-2,4-dibromo-9,10-anthracenedione $C_{14}H_7Br_2NO_2$ (1-amino-2,4-dibromoanthraquinone) [81-49-2] ZINRVIQBCHAZMM-UHFFFAOYSA-N | $5.5\times10^{7}$ | | HSDB (2015) | Q | 99 |
| 2,6-dibromo-4-cyanophenyl octanoate $C_{15}H_{17}BrNO_2$ [1689-99-2] DQKWXTIYGWPGOO-UHFFFAOYSA-N | $3.1\times10^{-1}$ $3.1\times10^{-1}$ $4.1\times10^{1}$ $<5.3$ | | Duchowicz et al. (2020) HSDB (2015) Duchowicz et al. (2020) Maniere et al. (2011) | V V Q ? | 186 12, 165 |
| bromobutide $C_{15}H_{22}BrNO$ [74712-19-9] WZDDLAZXUYIVMU-UHFFFAOYSA-N | $1.9\times10^{2}$ | | Ebert et al. (2023) | ? | 318 |
| (2E)-N,N'-bis(2,4,6-tribromophenyl)-2-butenediamide $C_{16}H_8Br_6N_2O_2$ [92484-07-6] IJUNKLAVMFKPCX-OWOJBTEDSA-N | $9.0\times10^{9}$ $5.1\times10^{8}$ $6.2\times10^{9}$ $7.2\times10^{13}$ | | Zhang et al. (2010) Zhang et al. (2010) Zhang et al. (2010) Zhang et al. (2010) | Q Q Q Q | 287, 288 287, 289 287, 290 287, 291 |
| SAYTEX BT 93 $C_{18}H_4Br_8N_2O_4$ [32588-76-4] DYIZJUDNMOIZQO-UHFFFAOYSA-N | $2.7\times10^{15}$ $2.7\times10^{15}$ $2.3\times10^{11}$ $3.5\times10^{9}$ $2.3\times10^{13}$ | | HSDB (2015) Zhang et al. (2010) Zhang et al. (2010) Zhang et al. (2010) Zhang et al. (2010) | Q Q Q Q Q | 99 287, 288 287, 289 287, 290 287, 291 |
| SAYTEX BN 451 $C_{20}H_{20}Br_4N_2O_4$ [52907-07-0] WOFYQUJNULCFLN-UHFFFAOYSA-N | $2.5\times10^{15}$ $1.4\times10^{11}$ $5.7\times10^{11}$ $1.6\times10^{15}$ | | Zhang et al. (2010) Zhang et al. (2010) Zhang et al. (2010) Zhang et al. (2010) | Q Q Q Q | 287, 288 287, 289 287, 290 287, 291 |





Table A7.2: Polybrominated diphenyl ethers (PBDEs) (...continued)

| Substance Formula (Trivial Name) [CAS Registry Number] InChIKey | $H_s^{cp}$ (at $T^\ominus$) $\left[\dfrac{\text{mol}}{\text{m}^3\,\text{Pa}}\right]$ | $\dfrac{\text{d}\ln H_s^{cp}}{\text{d}(1/T)}$ [K] | Reference | Type | Note |
|---|---|---|---|---|---|
| deltamethrin | 2.0 | | Duchowicz et al. (2020) | V | 186 |
| $C_{22}H_{19}Br_2NO_3$ | 2.0 | | HSDB (2015) | V | |
| [52918-63-5] | $4.0\times10^{-1}$ | | Mackay et al. (2006d) | V | |
| OWZREIFADZCYQD-NSHGMRRFSA-N | 2.0 | | Siebers and Mattusch (1996) | V | 12 |
| | $1.0\times10^3$ | | Duchowicz et al. (2020) | Q | |
| | $3.2\times10^1$ | | Maniere et al. (2011) | ? | 165 |
| 2,2'-(methylenedi-4,1-phenylene)bis(4,5,6,7-tetrabromo-1H-isoindole-1,3(2H)-dione | $5.3\times10^{14}$ | | Zhang et al. (2010) | Q | 287, 288 |
| $C_{29}H_{10}N_2O_4Br_8$ | $1.7\times10^{14}$ | | Zhang et al. (2010) | Q | 287, 289 |
| [32588-74-2] | $9.5\times10^9$ | | Zhang et al. (2010) | Q | 287, 290 |
| RHFKNNBDKARGKL-UHFFFAOYSA-N | $4.7\times10^{15}$ | | Zhang et al. (2010) | Q | 287, 291 |



### A7.3   Bromine, chlorine and fluorine (C, H, N, O, F, Cl, Br)

Table A7.3: Bromine, chlorine and fluorine (C, H, N, O, F, Cl, Br)

| Substance Formula (Trivial Name) [CAS Registry Number] InChIKey | $H_s^{cp}$ (at $T^\ominus$) $\left[\dfrac{\text{mol}}{\text{m}^3\,\text{Pa}}\right]$ | $\dfrac{\text{d}\ln H_s^{cp}}{\text{d}(1/T)}$ [K] | Reference | Type | Note |
|---|---|---|---|---|---|
| bromotrifluoromethane CF$_3$Br [75-63-8] RJCQBQGAPKAMLL-UHFFFAOYSA-N | $2.0\times10^{-5}$ | | Hine and Mookerjee (1975) | V | |
| | $2.0\times10^{-5}$ | | Yaws (2003) | X | 237 |
| | $2.0\times10^{-5}$ | | Irmann (1965) | C | |
| | $6.4\times10^{-6}$ | | Keshavarz et al. (2022) | Q | |
| | $5.3\times10^{-5}$ | | Duchowicz et al. (2020) | Q | 299 |
| | $4.7\times10^{-5}$ | | Gharagheizi et al. (2012) | Q | |
| | $1.8\times10^{-5}$ | | Gharagheizi et al. (2010) | Q | 246 |
| | $3.2\times10^{-5}$ | | Hilal et al. (2008) | Q | |
| | $1.1\times10^{-5}$ | | Modarresi et al. (2007) | Q | 67 |
| | $2.7\times10^{-5}$ | | Nirmalakhandan and Speece (1988) | Q | |
| | $5.6\times10^{-6}$ | | Irmann (1965) | Q | |
| | $2.0\times10^{-5}$ | | Duchowicz et al. (2020) | ? | 185, 21 |
| | $2.1\times10^{-5}$ | | Yaws (1999) | ? | 21 |
| dibromodifluoromethane CBr$_2$F$_2$ [75-61-6] AZSZCFSOHXEJQE-UHFFFAOYSA-N | $3.3\times10^{-4}$ | | HSDB (2015) | Q | 99 |
| 1-bromo-1,2,2,2-tetrafluoroethane C$_2$HBrF$_4$ (teflurane) [124-72-1] RZXZIZDRFQFCTA-UHFFFAOYSA-N | $1.7\times10^{-4}$ | 2700 | Allott et al. (1973) | L | |
| | $1.2\times10^{-4}$ | | Edelist et al. (1964) | M | 14 |
| | $1.4\times10^{-4}$ | | Keshavarz et al. (2022) | Q | |
| | $1.5\times10^{-3}$ | | Duchowicz et al. (2020) | Q | |
| | $2.1\times10^{-4}$ | | Hilal et al. (2008) | Q | |
| | $6.7\times10^{-5}$ | | Modarresi et al. (2007) | Q | 67 |
| | $1.7\times10^{-4}$ | | Duchowicz et al. (2020) | ? | 185, 21 |
| | $1.3\times10^{-4}$ | | Abraham and Weathersby (1994) | ? | 21 |
| | $1.7\times10^{-4}$ | | Abraham et al. (1990) | ? | |
| 1,2-dibromotetrafluoroethane C$_2$Br$_2$F$_4$ [124-73-2] KVBKAPANDHPRDG-UHFFFAOYSA-N | $2.7\times10^{-7}$ | | Duchowicz et al. (2020) | V | 186 |
| | $2.7\times10^{-7}$ | | HSDB (2015) | V | |
| | $2.4\times10^{-4}$ | | Duchowicz et al. (2020) | Q | |
| 1-bromo-2-fluorobenzene C$_6$H$_4$BrF [1072-85-1] IPWBFGUBXWMIPR-UHFFFAOYSA-N | $4.1\times10^{-3}$ | | Ebert et al. (2023) | ? | 316 |
| 1-bromo-3-fluorobenzene C$_6$H$_4$BrF [1073-06-9] QDFKKJYEIFBEFC-UHFFFAOYSA-N | $2.9\times10^{-3}$ | | Ebert et al. (2023) | ? | 316 |



Table A7.3: Bromine, chlorine and fluorine (C, H, N, O, F, Cl, Br) (... continued)

| Substance Formula (Trivial Name) [CAS Registry Number] InChIKey | $H_s^{cp}$ (at $T^{\ominus}$) $\left[\dfrac{\mathrm{mol}}{\mathrm{m^3\,Pa}}\right]$ | $\dfrac{\mathrm{d}\ln H_s^{cp}}{\mathrm{d}(1/T)}$ [K] | Reference | Type | Note |
|---|---|---|---|---|---|
| 1-bromo-4-fluorobenzene $C_6H_4BrF$ [460-00-4] AITNMTXHTIIIBB-UHFFFAOYSA-N | $5.3\times10^{-3}$ | 4400 | Hiatt (2013) | M | |
| bromopentafluorobenzene $C_6BrF_5$ [344-04-7] XEKTVXADUPBFOA-UHFFFAOYSA-N | $2.1\times10^{-3}$ $1.6\times10^{-4}$ $1.4\times10^{-4}$ $6.7\times10^{-4}$ | | Zhang et al. (2010) Zhang et al. (2010) Zhang et al. (2010) Zhang et al. (2010) | Q Q Q Q | 287, 288 287, 289 287, 290 287, 291 |
| halfenprox $C_{24}H_{23}BrF_2O_3$ [111872-58-3] WIFXJBMOTMKRMM-UHFFFAOYSA-N | $1.9\times10^{1}$ | | Ebert et al. (2023) | ? | 318 |
| bromethalin $C_{14}H_7Br_3F_3N_3O_4$ [63333-35-7] USMZPYXTVKAYST-UHFFFAOYSA-N | $2.5\times10^{3}$ | | HSDB (2015) | Q | 99 |
| brofluthrinate $C_{26}H_{22}BrF_2NO_4$ [160791-64-0] BUHNCQOJJZAOMJ-UHFFFAOYSA-N | $1.6\times10^{1}$ | | Ebert et al. (2023) | ? | 316 |
| bromochloromethane $CH_2BrCl$ [74-97-5] JPOXNPPZZKNXOV-UHFFFAOYSA-N | $6.6\times10^{-3}$ $7.8\times10^{-3}$ $6.8\times10^{-3}$ $5.8\times10^{-3}$ $5.8\times10^{-3}$ $6.2\times10^{-3}$ $1.1\times10^{-2}$ $2.5\times10^{-2}$ $5.7\times10^{-3}$ $5.3\times10^{-3}$ $6.2\times10^{-3}$ | 4700 4600 | Hiatt (2013) Kondoh and Nakajima (1997) HSDB (2015) Mackay et al. (2006b) Mackay et al. (1993) Yaws (2003) Gharagheizi et al. (2012) Gharagheizi et al. (2010) Yao et al. (2002) Katritzky et al. (1998) Yaws (1999) Fogg and Sangster (2003) | M M V V V X Q Q Q Q ? W | 237 246 229 21 791 |
| bromodichloromethane $CHCl_2Br$ [75-27-4] FMWLUWPQPKEARP-UHFFFAOYSA-N | $4.0\times10^{-3}$ $3.9\times10^{-3}$ $4.0\times10^{-3}$ $3.9\times10^{-3}$ $4.0\times10^{-3}$ $4.0\times10^{-3}$ $4.8\times10^{-3}$ $4.0\times10^{-3}$ $4.0\times10^{-3}$ $5.2\times10^{-3}$ $5.8\times10^{-3}$ | 5200 4900 5200 4900 5200 5200 3700 5200 5200 4700 | Burkholder et al. (2019) Burkholder et al. (2019) Burkholder et al. (2015) Burkholder et al. (2015) Sander et al. (2011) Sander et al. (2006) Fogg and Sangster (2003) Staudinger and Roberts (2001) Staudinger and Roberts (1996) Hiatt (2013) Ruiz-Bevia and Fernandez-Torres (2010) | L L L L L L L L L M M | 70 70 |



Table A7.3: Bromine, chlorine and fluorine (C, H, N, O, F, Cl, Br) (...continued)

| Substance Formula (Trivial Name) [CAS Registry Number] InChIKey | $H_s^{cp}$ (at $T^\ominus$) $\left[\dfrac{\text{mol}}{\text{m}^3\,\text{Pa}}\right]$ | $\dfrac{\text{d}\ln H_s^{cp}}{\text{d}(1/T)}$ [K] | Reference | Type | Note |
|---|---|---|---|---|---|
| | $2.9\times10^{-3}$ | | Zhang et al. (2002) | M | 14 |
| | $5.4\times10^{-3}$ | 4400 | Kondoh and Nakajima (1997) | M | |
| | $3.9\times10^{-3}$ | 4800 | Moore et al. (1995) | M | 792, 70 |
| | $4.8\times10^{-3}$ | 4200 | Tse et al. (1992) | M | |
| | $4.7\times10^{-3}$ | 5200 | Nicholson et al. (1984) | M | |
| | $3.5\times10^{-3}$ | 5200 | Ervin et al. (1980) | M | |
| | $4.7\times10^{-3}$ | | Warner et al. (1980) | M | |
| | $4.1\times10^{-3}$ | | Mackay et al. (2006b) | V | |
| | $4.1\times10^{-3}$ | | Mackay et al. (1993) | V | |
| | $4.6\times10^{-3}$ | 1200 | Goldstein (1982) | X | 298 |
| | $7.7\times10^{-3}$ | | Hilal et al. (2008) | C | |
| | $4.3\times10^{-3}$ | | Nicholson et al. (1984) | C | |
| | $4.7\times10^{-3}$ | | Nicholson et al. (1984) | C | 12 |
| | $4.7\times10^{-3}$ | | Shen (1982) | C | |
| | $8.8\times10^{-3}$ | | Keshavarz et al. (2022) | Q | |
| | $9.4\times10^{-3}$ | | Duchowicz et al. (2020) | Q | 184 |
| | $3.1\times10^{-3}$ | | Raventos-Duran et al. (2010) | Q | 271, 243 |
| | $3.9\times10^{-3}$ | | Raventos-Duran et al. (2010) | Q | 244 |
| | $9.9\times10^{-3}$ | | Raventos-Duran et al. (2010) | Q | 245 |
| | $3.9\times10^{-3}$ | | Hilal et al. (2008) | Q | |
| | $2.4\times10^{-3}$ | | Modarresi et al. (2007) | Q | 67 |
| | | 4100 | Kühne et al. (2005) | Q | |
| | $4.8\times10^{-3}$ | | Yaffe et al. (2003) | Q | 248, 249 |
| | $3.2\times10^{-3}$ | | Katritzky et al. (1998) | Q | |
| | $4.7\times10^{-3}$ | | Duchowicz et al. (2020) | ? | 185, 21 |
| | $6.2\times10^{-3}$ | | Mackay et al. (2006b) | ? | |
| | | 3800 | Kühne et al. (2005) | ? | |
| | $6.2\times10^{-3}$ | | Mackay et al. (1993) | ? | |
| bromotrichloromethane CBrCl$_3$ [75-62-7] XNNQFQFUQLJSQT-UHFFFAOYSA-N | $2.7\times10^{-2}$ | | HSDB (2015) | Q | 99 |
| dibromochloromethane CHClBr$_2$ [124-48-1] GATVIKZLVQHOMN-UHFFFAOYSA-N | $8.6\times10^{-3}$ | 5500 | Burkholder et al. (2019) | L | |
| | $7.2\times10^{-3}$ | 5200 | Burkholder et al. (2019) | L | 70 |
| | $8.6\times10^{-3}$ | 5500 | Burkholder et al. (2015) | L | |
| | $7.2\times10^{-3}$ | 5200 | Burkholder et al. (2015) | L | 70 |
| | $8.6\times10^{-3}$ | 5500 | Sander et al. (2011) | L | |
| | $8.6\times10^{-3}$ | 5500 | Sander et al. (2006) | L | |
| | $8.7\times10^{-3}$ | 4400 | Fogg and Sangster (2003) | L | |
| | $8.6\times10^{-3}$ | 5500 | Staudinger and Roberts (2001) | L | |
| | $8.5\times10^{-3}$ | 5500 | Staudinger and Roberts (1996) | L | |
| | $1.1\times10^{-2}$ | 5300 | Hiatt (2013) | M | |
| | $1.1\times10^{-2}$ | | Ruiz-Bevia and Fernandez-Torres (2010) | M | |
| | $4.6\times10^{-3}$ | | Zhang et al. (2002) | M | 14 |
| | $9.8\times10^{-3}$ | 5100 | Kondoh and Nakajima (1997) | M | |





Table A7.3: Bromine, chlorine and fluorine (C, H, N, O, F, Cl, Br) (. . . continued)

| Substance Formula (Trivial Name) [CAS Registry Number] InChIKey | $H_s^{cp}$ (at $T^{\ominus}$) $\left[\dfrac{\text{mol}}{\text{m}^3\,\text{Pa}}\right]$ | $\dfrac{\text{d}\ln H_s^{cp}}{\text{d}(1/T)}$ [K] | Reference | Type | Note |
|---|---|---|---|---|---|
| | $7.4 \times 10^{-3}$ | 4600 | Moore et al. (1995) | M | 793, 70 |
| | $9.3 \times 10^{-3}$ | 4600 | Tse et al. (1992) | M | |
| | $8.5 \times 10^{-3}$ | 6400 | Ashworth et al. (1988) | M | 278 |
| | $8.6 \times 10^{-3}$ | 5200 | Nicholson et al. (1984) | M | |
| | $8.5 \times 10^{-3}$ | 5000 | Ervin et al. (1980) | M | |
| | $1.3 \times 10^{-2}$ | | Warner et al. (1980) | M | |
| | $1.2 \times 10^{-2}$ | | Mackay et al. (2006b) | V | |
| | $1.2 \times 10^{-2}$ | | Goldstein (1982) | X | 446 |
| | $1.2 \times 10^{-2}$ | 2500 | Goldstein (1982) | X | 298 |
| | $1.2 \times 10^{-2}$ | | Nicholson et al. (1984) | C | |
| | $1.1 \times 10^{-2}$ | | Nicholson et al. (1984) | C | 12 |
| | $1.3 \times 10^{-2}$ | | Shen (1982) | C | |
| | $5.4 \times 10^{-3}$ | | Hilal et al. (2008) | Q | |
| | $2.2 \times 10^{-3}$ | | Modarresi et al. (2007) | Q | 67 |
| | | 4800 | Kühne et al. (2005) | Q | |
| | $1.3 \times 10^{-2}$ | | Yaffe et al. (2003) | Q | 248, 249 |
| | $1.6 \times 10^{-2}$ | | Katritzky et al. (1998) | Q | |
| | | 4600 | Kühne et al. (2005) | ? | |
| | $1.2 \times 10^{-2}$ | | Mackay et al. (1993) | ? | |
| 1-chloro-2-bromoethane $C_2H_4BrCl$ [107-04-0] IBYHHJPAARCAIE-UHFFFAOYSA-N | $1.1 \times 10^{-2}$ | | Hine and Mookerjee (1975) | V | |
| | $1.1 \times 10^{-2}$ | | Sieg et al. (2008) | C | |
| | $1.2 \times 10^{-2}$ | | Keshavarz et al. (2022) | Q | |
| | $3.6 \times 10^{-3}$ | | Duchowicz et al. (2020) | Q | |
| | $1.8 \times 10^{-2}$ | | Hilal et al. (2008) | Q | |
| | $5.5 \times 10^{-3}$ | | Modarresi et al. (2007) | Q | 67 |
| | $1.2 \times 10^{-2}$ | | Yaffe et al. (2003) | Q | 248, 249 |
| | $1.7 \times 10^{-2}$ | | Katritzky et al. (1998) | Q | |
| | $3.7 \times 10^{-3}$ | | Nirmalakhandan and Speece (1988) | Q | |
| | $1.1 \times 10^{-2}$ | | Duchowicz et al. (2020) | ? | 185, 21 |
| 1,2-dibromo-1,1-dichloroethane $C_2H_2Br_2Cl_2$ [75-81-0] FIYBYNHDEOSJPL-UHFFFAOYSA-N | $6.2 \times 10^{-2}$ | | HSDB (2015) | Q | 99 |
| 1-bromo-3-chloropropane $C_3H_6BrCl$ [109-70-6] MFESCIUQSIBMSM-UHFFFAOYSA-N | $3.9 \times 10^{-2}$ | | HSDB (2015) | Q | 99 |
| 1,2-dibromo-3-chloropropane $C_3H_5Br_2Cl$ [96-12-8] WBEJYOJJBDISQU-UHFFFAOYSA-N | $9.7 \times 10^{-2}$ | 7100 | Hiatt (2013) | M | |
| | $5.0 \times 10^{-1}$ | 10000 | Kondoh and Nakajima (1997) | M | |
| | $6.6 \times 10^{-2}$ | | HSDB (2015) | V | |
| | $6.7 \times 10^{-2}$ | | Meylan and Howard (1991) | V | |
| | $9.0 \times 10^{-2}$ | | Hilal et al. (2008) | Q | |
| | $1.1 \times 10^{-2}$ | | Modarresi et al. (2007) | Q | 67 |
| | $6.4 \times 10^{-2}$ | | Yaffe et al. (2003) | Q | 248, 249 |
| | $2.1 \times 10^{-1}$ | | Katritzky et al. (1998) | Q | |





Table A7.3: Bromine, chlorine and fluorine (C, H, N, O, F, Cl, Br) (...continued)

| Substance<br>Formula<br>(Trivial Name)<br>[CAS Registry Number]<br>InChIKey | $H_s^{cp}$<br>(at $T^{\ominus}$)<br>$\left[\dfrac{\text{mol}}{\text{m}^3\,\text{Pa}}\right]$ | $\dfrac{\mathrm{d}\ln H_s^{cp}}{\mathrm{d}(1/T)}$<br><br>[K] | Reference | Type | Note |
|---|---|---|---|---|---|
| | $1.6\times10^{-2}$ | | Meylan and Howard (1991) | Q | |
| | $4.0\times10^{-2}$ | | MacBean (2012a) | ? | |
| 1,2,3,4,5-pentabromo-6-<br>chlorocyclohexane | $1.0\times10^{1}$ | | Zhang et al. (2010) | Q | 287, 288 |
| $C_6H_6Br_5Cl$ | $1.1\times10^{2}$ | | Zhang et al. (2010) | Q | 287, 289 |
| [87-84-3] | $1.8\times10^{3}$ | | Zhang et al. (2010) | Q | 287, 290 |
| UZOSVZSBPTTWIG-UHFFFAOYSA-N | $1.2\times10^{1}$ | | Zhang et al. (2010) | Q | 287, 291 |
| 1,2,3,4-tetrabromo-5,6-<br>dichlorocyclohexane | 3.4 | | Zhang et al. (2010) | Q | 287, 288 |
| $C_6H_6Br_4Cl_2$ | $6.2\times10^{1}$ | | Zhang et al. (2010) | Q | 287, 289 |
| GABFTOZBVQIDBB-UHFFFAOYSA-N | $9.9\times10^{2}$ | | Zhang et al. (2010) | Q | 287, 290 |
| | 6.2 | | Zhang et al. (2010) | Q | 287, 291 |
| 1,2,3-tribromo-4,5,6-<br>trichlorocyclohexane | 1.1 | | Zhang et al. (2010) | Q | 287, 288 |
| $C_6H_6Br_3Cl_3$ | $3.6\times10^{1}$ | | Zhang et al. (2010) | Q | 287, 289 |
| ZDEQCIGFZLZXBZ-UHFFFAOYSA-N | $4.1\times10^{2}$ | | Zhang et al. (2010) | Q | 287, 290 |
| | 3.0 | | Zhang et al. (2010) | Q | 287, 291 |
| 1-bromo-4-chlorobenzene | $6.8\times10^{-3}$ | | Duchowicz et al. (2020) | V | 186 |
| $C_6H_4BrCl$ | $6.8\times10^{-3}$ | | Mackay and Shiu (1981) | V | |
| [106-39-8] | $7.7\times10^{-3}$ | | Duchowicz et al. (2020) | Q | |
| NHDODQWIKUYWMW-UHFFFAOYSA-N | $7.8\times10^{-3}$ | | Raventos-Duran et al. (2010) | Q | 271, 243 |
| | $1.2\times10^{-2}$ | | Raventos-Duran et al. (2010) | Q | 244 |
| | $6.2\times10^{-3}$ | | Raventos-Duran et al. (2010) | Q | 245 |
| | $9.0\times10^{-3}$ | | Hilal et al. (2008) | Q | |
| | $6.8\times10^{-3}$ | | Modarresi et al. (2007) | Q | 67 |
| | $6.9\times10^{-3}$ | | Yaffe et al. (2003) | Q | 248, 249 |
| | $3.3\times10^{-2}$ | | Katritzky et al. (1998) | Q | |
| 1,2,4-tribromo-3,5,6-<br>trichlorobenzene<br>$C_6Br_3Cl_3$<br>[13075-01-9]<br>XAIKFWSNTFEXDC-UHFFFAOYSA-N | $4.1\times10^{-2}$ | | HSDB (2015) | Q | 99 |
| 1-(bromomethyl)-2-chlorobenzene<br>$C_7H_6BrCl$<br>[611-17-6]<br>PURSZYWBIQIANP-UHFFFAOYSA-N | $1.9\times10^{-2}$ | | HSDB (2015) | Q | 447 |
| 2-bromo-4-chloro-1-<br>methoxybenzene<br>$C_7H_6BrClO$<br>(2-bromo-4-chloroanisole)<br>[60633-25-2]<br>YJEMGEBDXDPBSP-UHFFFAOYSA-N | $1.8\times10^{-2}$ | | Pfeifer et al. (2001) | M | 731 |



Table A7.3: Bromine, chlorine and fluorine (C, H, N, O, F, Cl, Br) (...continued)

| Substance Formula (Trivial Name) [CAS Registry Number] InChIKey | $H_s^{cp}$ (at $T^\ominus$) $\left[\dfrac{\text{mol}}{\text{m}^3\,\text{Pa}}\right]$ | $\dfrac{\text{d}\ln H_s^{cp}}{\text{d}(1/T)}$ [K] | Reference | Type | Note |
|---|---|---|---|---|---|
| 2-bromo-6-chloro-1-methoxybenzene C$_7$H$_6$BrClO (2-bromo-6-chloroanisole) [174913-10-1] NWOYYECMNBWCNK-UHFFFAOYSA-N | $1.4\times10^{-2}$ | | Pfeifer et al. (2001) | M | 731 |
| 4-bromo-2-chloro-1-methoxybenzene C$_7$H$_6$BrClO (4-bromo-2-chloroanisole) [50638-47-6] FPIQNBOUYZLESW-UHFFFAOYSA-N | $1.3\times10^{-2}$ | | Pfeifer et al. (2001) | M | 731 |
| 2-bromo-3,5-dichloro-1-methoxybenzene C$_7$H$_5$BrCl$_2$O (2-bromo-3,5-dichloroanisole) [73931-43-8] LCHJNNXHVDDVAT-UHFFFAOYSA-N | $1.1\times10^{-2}$ | | Pfeifer et al. (2001) | M | 731 |
| 2-bromo-4,6-dichloro-1-methoxybenzene C$_7$H$_5$BrCl$_2$O (2-bromo-4,6-dichloroanisole) [60633-26-3] OEYKUHBCPJRXGZ-UHFFFAOYSA-N | $8.2\times10^{-3}$ $1.2\times10^{-2}$ | 3100 | Diaz et al. (2005) Pfeifer et al. (2001) | M M | 731 |
| 4-bromo-2,3-dichloro-1-methoxybenzene C$_7$H$_5$BrCl$_2$O (4-bromo-2,3-dichloroanisole) [109803-52-3] BTRCDLZQISXWHZ-UHFFFAOYSA-N | $1.1\times10^{-2}$ | | Pfeifer et al. (2001) | M | 731 |
| 4-bromo-2,6-dichloro-1-methoxybenzene C$_7$H$_5$BrCl$_2$O (4-bromo-2,6-dichloroanisole) [19240-91-6] OAYSFAKCFYRCRU-UHFFFAOYSA-N | $1.2\times10^{-2}$ $1.2\times10^{-2}$ | 4900 | Diaz et al. (2005) Pfeifer et al. (2001) | M M | 731 |
| 4-bromo-3,5-dichloro-1-methoxybenzene C$_7$H$_5$BrCl$_2$O (4-bromo-3,5-dichloroanisole) [174913-20-3] SVEZPKVUIXAEOX-UHFFFAOYSA-N | $1.1\times10^{-2}$ | | Pfeifer et al. (2001) | M | 731 |



Table A7.3: Bromine, chlorine and fluorine (C, H, N, O, F, Cl, Br) (...continued)

| Substance Formula (Trivial Name) [CAS Registry Number] InChIKey | $H_s^{cp}$ (at $T^{\ominus}$) $\left[\dfrac{\text{mol}}{\text{m}^3\,\text{Pa}}\right]$ | $\dfrac{\text{d}\ln H_s^{cp}}{\text{d}(1/T)}$ [K] | Reference | Type | Note |
|---|---|---|---|---|---|
| 3-bromo-2,4-dichloro-1-methoxybenzene $C_7H_5BrCl_2O$ (3-bromo-2,4-dichloroanisole) [174913-16-7] OXINPOQPENRUHL-UHFFFAOYSA-N | $1.2\times10^{-2}$ | | Ebert et al. (2023) | ? | 789 |
| 3-bromo-2,6-dichloro-1-methoxybenzene $C_7H_5BrCl_2O$ (3-bromo-2,6-dichloroanisole) [174913-18-9] ANLVZYYGYBXMIX-UHFFFAOYSA-N | $1.1\times10^{-2}$ | 2700 | Diaz et al. (2005) | M | 790 |
| 5-bromo-2,4-dichloro-1-methoxybenzene $C_7H_5BrCl_2O$ (5-bromo-2,4-dichloroanisole) [174913-22-5] SJIIFORJBWPNPM-UHFFFAOYSA-N | $1.1\times10^{-2}$ | | Pfeifer et al. (2001) | M | 731 |
| 6-bromo-2,3-dichloro-1-methoxybenzene $C_7H_5BrCl_2O$ (6-bromo-2,3-dichloroanisole) [174913-23-6] PNLDDSLZYVNPQW-UHFFFAOYSA-N | $1.1\times10^{-2}$ | | Pfeifer et al. (2001) | M | 731 |
| 6-bromo-2,5-dichloro-1-methoxybenzene $C_7H_5BrCl_2O$ (6-bromo-2,5-dichloroanisole) [174913-14-5] BVSOVPUWHJWGEP-UHFFFAOYSA-N | $7.7\times10^{-3}$ | 3000 | Diaz et al. (2005) | M | |
| 2-bromo-3,4,5-trichloro-1-methoxybenzene $C_7H_4BrCl_3O$ (2-bromo-3,4,5-trichloroanisole) JIVCPRHFBFVZQL-UHFFFAOYSA-N | $9.8\times10^{-3}$ | | Pfeifer et al. (2001) | M | 731 |
| 3-bromo-2,4,6-trichloro-1-methoxybenzene $C_7H_4BrCl_3O$ (3-bromo-2,4,6-trichloroanisole) [174913-28-1] OIHHRLYOSUNYTC-UHFFFAOYSA-N | $1.0\times10^{-2}$ | | Pfeifer et al. (2001) | M | 731 |



Table A7.3: Bromine, chlorine and fluorine (C, H, N, O, F, Cl, Br) (... continued)

| Substance Formula (Trivial Name) [CAS Registry Number] InChIKey | $H_s^{cp}$ (at $T^{\ominus}$) $\left[\dfrac{\mathrm{mol}}{\mathrm{m^3\,Pa}}\right]$ | $\dfrac{\mathrm{d}\ln H_s^{cp}}{\mathrm{d}(1/T)}$ [K] | Reference | Type | Note |
|---|---|---|---|---|---|
| 3-bromo-2,5,6-trichloro-1-methoxybenzene $C_7H_4BrCl_3O$ (3-bromo-2,5,6-trichloroanisole) [78647-93-5] ZUXVGCIQOSXRJN-UHFFFAOYSA-N | $1.0\times10^{-2}$ | | Pfeifer et al. (2001) | M | 731 |
| 4-bromo-2,3,6-trichloro-1-methoxybenzene $C_7H_4BrCl_3O$ (4-bromo-2,3,6-trichloroanisole) [78647-87-7] VZRSUQWPXPUWCS-UHFFFAOYSA-N | $1.0\times10^{-2}$ | | Pfeifer et al. (2001) | M | 731 |
| 6-bromo-2,3,4-trichloro-1-methoxybenzene $C_7H_4BrCl_3O$ (6-bromo-2,3,4-trichloroanisole) GVCDCZPXMGMNTR-UHFFFAOYSA-N | $1.1\times10^{-2}$ | | Pfeifer et al. (2001) | M | 731 |
| 4-bromo-2,3,5,6-tetrachloro-1-methoxybenzene $C_7H_3BrCl_4O$ (4-bromo-2,3,5,6-tetrachloroanisole) [174913-33-8] YSVAURKXNCNUNS-UHFFFAOYSA-N | $9.2\times10^{-3}$ | | Pfeifer et al. (2001) | M | 731 |
| 2,4-dibromo-3-chloro-1-methoxybenzene $C_7H_5Br_2ClO$ (2,4-dibromo-3-chloroanisole) YXYRJAVLYYNRCE-UHFFFAOYSA-N | $1.0\times10^{-2}$ | | Ebert et al. (2023) | ? | 789 |
| 2,4-dibromo-5-chloro-1-methoxybenzene $C_7H_5Br_2ClO$ (2,4-dibromo-5-chloroanisole) [174913-38-3] OTZRCEAVEJYZQS-UHFFFAOYSA-N | $1.0\times10^{-2}$ | | Ebert et al. (2023) | ? | 789 |
| 2,6-dibromo-3-chloro-1-methoxybenzene $C_7H_5Br_2ClO$ (2,6-dibromo-3-chloroanisole) AHBGIHFKTBTQJP-UHFFFAOYSA-N | $7.4\times10^{-3}$ | 770 | Diaz et al. (2005) | M | 790 |



Table A7.3: Bromine, chlorine and fluorine (C, H, N, O, F, Cl, Br) (...continued)

| Substance Formula (Trivial Name) [CAS Registry Number] InChIKey | $H_s^{cp}$ (at $T^{\ominus}$) $\left[\dfrac{\mathrm{mol}}{\mathrm{m^3\,Pa}}\right]$ | $\dfrac{\mathrm{d\ln} H_s^{cp}}{\mathrm{d}(1/T)}$ [K] | Reference | Type | Note |
|---|---|---|---|---|---|
| 2,6-dibromo-4-chloro-1-methoxybenzene C$_7$H$_5$Br$_2$ClO (2,6-dibromo-4-chloroanisole) [174913-44-1] WHRDPXDGAMJNGH-UHFFFAOYSA-N | $2.0\times10^{-2}$ $1.1\times10^{-2}$ | 6700 | Diaz et al. (2005) Pfeifer et al. (2001) | M M | 731 |
| 2,4-dibromo-3,5-dichloro-1-methoxybenzene C$_7$H$_4$Br$_2$Cl$_2$O (2,4-dibromo-3,5-dichloroanisole) [174913-52-1] YZMXEWDTXSTFIJ-UHFFFAOYSA-N | $9.1\times10^{-3}$ | | Pfeifer et al. (2001) | M | 731 |
| 2,4-dibromo-5,6-dichloro-1-methoxybenzene C$_7$H$_4$Br$_2$Cl$_2$O (2,4-dibromo-5,6-dichloroanisole) MUUDTVDSRYEMBU-UHFFFAOYSA-N | $9.8\times10^{-3}$ | | Pfeifer et al. (2001) | M | 731 |
| 2,3-dibromo-5,6-dichloro-1-methoxybenzene C$_7$H$_4$Br$_2$Cl$_2$O (2,3-dibromo-5,6-dichloroanisole) WDRCOFFEZVLCLX-UHFFFAOYSA-N | $9.1\times10^{-3}$ | | Pfeifer et al. (2001) | M | 731 |
| 2,6-dibromo-3,4,5-trichloro-1-methoxybenzene C$_7$H$_3$Br$_2$Cl$_3$O (2,6-dibromo-3,4,5-trichloroanisole) RFUGYJHXLQAXOX-UHFFFAOYSA-N | $8.6\times10^{-3}$ | | Pfeifer et al. (2001) | M | 731 |
| 2,4,6-tribromo-3-chloro-1-methoxybenzene C$_7$H$_4$Br$_3$ClO (2,4,6-tribromo-3-chloroanisole) [174913-78-1] FWKUBDOTUFVPBX-UHFFFAOYSA-N | $9.1\times10^{-3}$ | | Pfeifer et al. (2001) | M | 731 |
| 2,3-dibromo-7,8-dichlorodibenzo-$p$-dioxin C$_{12}$H$_4$Br$_2$Cl$_2$O$_2$ [50585-40-5] SBSJXPIUKNBZND-UHFFFAOYSA-N | 3.2 | | Ebert et al. (2023) | ? | 318 |



Table A7.3: Bromine, chlorine and fluorine (C, H, N, O, F, Cl, Br) (. . . continued)

| Substance Formula (Trivial Name) [CAS Registry Number] InChIKey | $H_s^{cp}$ (at $T^{\ominus}$) $\left[\dfrac{\mathrm{mol}}{\mathrm{m^3\,Pa}}\right]$ | $\dfrac{\mathrm{d}\ln H_s^{cp}}{\mathrm{d}(1/T)}$ [K] | Reference | Type | Note |
|---|---|---|---|---|---|
| 2',4',5',7'-tetrabromo-3,4,5,6-tetrachlorofluorescein $C_{20}H_4Br_4Cl_4O_5$ [13473-26-2] ZYIBVBKZZZDFOY-UHFFFAOYSA-N | $1.5\times10^{13}$ $1.9\times10^8$ $8.0\times10^{10}$ $2.2\times10^8$ | | Zhang et al. (2010) Zhang et al. (2010) Zhang et al. (2010) Zhang et al. (2010) | Q Q Q Q | 287, 288 287, 289 287, 290 287, 291 |
| bromochloroacetonitrile $C_2HBrClN$ [83463-62-1] BMWPPNAUMLRKML-UHFFFAOYSA-N | 8.2 | | HSDB (2015) | Q | 99 |
| N,N'-dimethyl-3,3',4-tribromo-4,5,5'-trichloro-2,2'-bipyrrole $C_{10}H_6Br_3Cl_3N_2$ (DBP-Br3Cl3a) [400766-93-0] JSQYFPHYSOZLPT-UHFFFAOYSA-N | 7.1 9.5 | | Tittlemier et al. (2004) Hilal et al. (2008) | V Q | |
| N,N'-dimethyl- 3,4,4'-tribromo-3',5,5'-trichloro-2,2'-bipyrrole $C_{10}H_6Br_3Cl_3N_2$ (DBP-Br3Cl3b) [666856-68-4] GUWAXPTVCYBOMV-UHFFFAOYSA-N | $3.3\times10^1$ 9.5 | | Tittlemier et al. (2004) Hilal et al. (2008) | V Q | |
| N,N'-dimethyl-3,3',4,4'-tetrabromo-5,5'-dichloro-2,2'-bipyrrole $C_{10}H_6Br_4Cl_2N_2$ (DBP-Br4Cl2) [253798-64-0] FQGSQKFVAAJWFY-UHFFFAOYSA-N | $2.8\times10^1$ $1.8\times10^1$ | | Tittlemier et al. (2004) Hilal et al. (2008) | V Q | |
| N,N'-dimethyl-3,3',4,4',5-pentabromo-5'-chloro-2,2'-bipyrrole $C_{10}H_6Br_5ClN_2$ (DBP-Br5Cl) [400767-00-2] LIYSVIAQQQEQBT-UHFFFAOYSA-N | $1.5\times10^2$ $3.0\times10^1$ | | Tittlemier et al. (2004) Hilal et al. (2008) | V Q | |
| 1-bromo-3-chloro-5,5-dimethylhydantoin $C_5H_6BrClN_2O_2$ [16079-88-2] PIEXCQIOSMOEOU-UHFFFAOYSA-N | $1.2\times10^1$ | | HSDB (2015) | Q | 99 |



Table A7.3: Bromine, chlorine and fluorine (C, H, N, O, F, Cl, Br) (. . . continued)

| Substance Formula (Trivial Name) [CAS Registry Number] InChIKey | $H_s^{cp}$ (at $T^{\ominus}$) $\left[\dfrac{\mathrm{mol}}{\mathrm{m^3\,Pa}}\right]$ | $\dfrac{\mathrm{d}\ln H_s^{cp}}{\mathrm{d}(1/T)}$ [K] | Reference | Type | Note |
|---|---|---|---|---|---|
| 3-(4-bromo-3-chlorophenyl)-1-methoxy-1-methylurea | $2.2\times10^3$ | | HSDB (2015) | V | |
| $C_9H_{10}BrClN_2O_2$ | $3.2\times10^3$ | | Mackay et al. (2006d) | V | |
| (chlorbromuron) | $2.5\times10^3$ | | MacBean (2012a) | ? | |
| [13360-45-7] | | | | | |
| NLYNUTMZTCLNOO-UHFFFAOYSA-N | | | | | |
| N-(4-bromo-2,6-dichloro-3-methylphenyl)acetamide | $6.7\times10^3$ | | Zhang et al. (2010) | Q | 287, 288 |
| $C_9H_8BrCl_2NO$ | $6.2\times10^2$ | | Zhang et al. (2010) | Q | 287, 289 |
| [68399-95-1] | $2.1\times10^4$ | | Zhang et al. (2010) | Q | 287, 290 |
| MRPNWLPYJSSISQ-UHFFFAOYSA-N | $6.5\times10^3$ | | Zhang et al. (2010) | Q | 287, 291 |
| halacrinate | $2.4\times10^2$ | | MacBean (2012a) | ? | |
| $C_{12}H_7NO_2BrCl$ | | | | | |
| [34462-96-9] | | | | | |
| YDNLKBDXQCHOTH-UHFFFAOYSA-N | | | | | |
| bromuconazole | $3.3\times10^4$ | | Duchowicz et al. (2020) | V | 186 |
| $C_{13}H_{12}BrCl_2N_3O$ | $1.2\times10^5$ | | HSDB (2015) | V | |
| [116255-48-2] | $1.4\times10^2$ | | Duchowicz et al. (2020) | Q | |
| HJJVPARKXDDIQD-UHFFFAOYSA-N | $1.1\times10^5$ | | Maniere et al. (2011) | ? | 12, 165 |
| $(\pm)$-$(2R^*,4S^*)$-bromuconazole | $6.4\times10^4$ | | Ebert et al. (2023) | ? | 365 |
| $C_{13}H_{12}BrCl_2N_3O$ | | | | | |
| [114544-80-8] | | | | | |
| $(\pm)$-$(2R^*,4R^*)$-bromuconazole | $9.5\times10^4$ | | Ebert et al. (2023) | ? | 365 |
| $C_{13}H_{12}BrCl_2N_3O$ | | | | | |
| [114544-81-9] | | | | | |
| 5,7-dibromo-2-(5-bromo-7-chloro-1,3-dihydro-3-oxo-2H-indol-2-ylidene)-1,2-dihydro-3H-indol-3-one | $4.2\times10^9$ | | Zhang et al. (2010) | Q | 287, 288 |
| $C_{16}H_6Br_3ClN_2O_2$ | $3.3\times10^{15}$ | | Zhang et al. (2010) | Q | 287, 289 |
| [85702-64-3] | $2.4\times10^5$ | | Zhang et al. (2010) | Q | 287, 290 |
| UOWUGDPSFZLTMZ-BUHFOSPRSA-N | $8.8\times10^9$ | | Zhang et al. (2010) | Q | 287, 291 |
| chlorantraniliprole | $7.0\times10^{15}$ | | HSDB (2015) | Q | 99 |
| $C_{18}H_{14}BrCl_2N_5O_2$ | $3.1\times10^8$ | | Maniere et al. (2011) | ? | 241, 165 |
| [500008-45-7] | | | | | |
| IZCBNTYXTMZSDP-UHFFFAOYSA-N | | | | | |
| cyantraniliprole | $5.9\times10^{12}$ | | Maniere et al. (2011) | ? | 241, 165 |
| $C_{19}H_{14}BrClN_6O_2$ | | | | | |
| [736994-63-1] | | | | | |
| DVBUIBGJRQBEDP-UHFFFAOYSA-N | | | | | |



Table A7.3: Bromine, chlorine and fluorine (C, H, N, O, F, Cl, Br) (... continued)

| Substance Formula (Trivial Name) [CAS Registry Number] InChIKey | $H_s^{cp}$ (at $T^{\ominus}$) $\left[\dfrac{\text{mol}}{\text{m}^3\,\text{Pa}}\right]$ | $\dfrac{\text{d}\ln H_s^{cp}}{\text{d}(1/T)}$ [K] | Reference | Type | Note |
|---|---|---|---|---|---|
| tribromofluoromethane CBr$_3$F [353-54-8] IHZAEIHJPNTART-UHFFFAOYSA-N | $1.5 \times 10^{-3}$ | | Fogg and Sangster (2003) | V | |
| bromochlorodifluoromethane CBrClF$_2$ [353-59-3] MEXUFEQDCXZEON-UHFFFAOYSA-N | $6.1 \times 10^{-5}$ $1.0 \times 10^{-4}$ $2.6 \times 10^{-4}$ $6.4 \times 10^{-5}$ $8.6 \times 10^{-5}$ $6.0 \times 10^{-5}$ | | Yaws (2003) HSDB (2015) Gharagheizi et al. (2012) Gharagheizi et al. (2010) Hilal et al. (2008) Yaws (1999) | X Q Q Q Q ? | 237, 153 99 246 21, 153 |
| 1-bromo-1-chloro-2,2,2-trifluoroethane C$_2$HBrClF$_3$ (halothane) [151-67-7] BCQZXOMGPXTTIC-UHFFFAOYSA-N | $5.6 \times 10^{-4}$ $3.1 \times 10^{-4}$ $5.3 \times 10^{-4}$ $2.8 \times 10^{-4}$ $3.3 \times 10^{-4}$ $5.3 \times 10^{-4}$ $3.2 \times 10^{-4}$ $3.4 \times 10^{-4}$ $4.9 \times 10^{-4}$ $4.2 \times 10^{-4}$ $2.7 \times 10^{-3}$ $4.8 \times 10^{-4}$ $8.8 \times 10^{-4}$ $1.7 \times 10^{-4}$ $1.0 \times 10^{-3}$ $4.9 \times 10^{-4}$ $4.9 \times 10^{-4}$ $4.7 \times 10^{-4}$ $3.4 \times 10^{-4}$ $4.8 \times 10^{-4}$ | 4700 4200 5000 4100 5000 | Fogg and Sangster (2003) Steward et al. (1973) Allott et al. (1973) Guitart et al. (1989) Lerman et al. (1983) Smith et al. (1981b) Stoelting and Longshore (1972) Saidman et al. (1966) Yaws (2003) Keshavarz et al. (2022) Duchowicz et al. (2020) Gharagheizi et al. (2010) Hilal et al. (2008) Modarresi et al. (2007) Kühne et al. (2005) English and Carroll (2001) Duchowicz et al. (2020) HSDB (2015) Kühne et al. (2005) Yaws (1999) Abraham and Weathersby (1994) Abraham et al. (1990) | L L L M M M M M X Q Q Q Q Q Q Q ? ? ? ? ? ? | 14 14 14 14 237, 79 184 246 67 230, 231 185, 21 419 21, 79 21 |
| tralopyril C$_{12}$H$_5$BrClF$_3$N$_2$ [122454-29-9] XNFIRYXKTXAHAC-UHFFFAOYSA-N | $4.5 \times 10^3$ | | Ebert et al. (2023) | ? | 316 |
| chlorfenapyr C$_{15}$H$_{11}$BrClF$_3$N$_2$O [122453-73-0] CWFOCCVIPCEQCK-UHFFFAOYSA-N | $1.7 \times 10^3$ | | HSDB (2015) | Q | 99 |



Table A7.3: Bromine, chlorine and fluorine (C, H, N, O, F, Cl, Br) (...continued)

| Substance<br>Formula<br>(Trivial Name)<br>[CAS Registry Number]<br>InChIKey | $H_s^{cp}$<br>(at $T^{\ominus}$)<br>$\left[\dfrac{\text{mol}}{\text{m}^3\,\text{Pa}}\right]$ | $\dfrac{\mathrm{d}\ln H_s^{cp}}{\mathrm{d}(1/T)}$<br><br>[K] | Reference | Type | Note |
|---|---|---|---|---|---|
| fluazolate<br>$C_{15}H_{12}N_2O_2BrClF_4$<br>[174514-07-9]<br>FKLQIONHGSFYJY-UHFFFAOYSA-N | $1.3\times10^1$ | | MacBean (2012a) | ? | |



## A8   Organic species with iodine (I)

### A8.1   Iodocarbons (C, H, O, Cl, I)

Table A8.1: Iodocarbons (C, H, O, Cl, I)

| Substance<br>Formula<br>(Trivial Name)<br>[CAS Registry Number]<br>InChIKey | $H_s^{cp}$<br>(at $T^{\ominus}$)<br>$\left[\dfrac{\text{mol}}{\text{m}^3\,\text{Pa}}\right]$ | $\dfrac{\text{d}\ln H_s^{cp}}{\text{d}(1/T)}$<br><br>[K] | Reference | Type | Note |
|---|---|---|---|---|---|
| iodomethane | $2.0\times10^{-3}$ | 3200 | Burkholder et al. (2019) | L | 1 |
| $CH_3I$ | $1.5\times10^{-3}$ | 3900 | Burkholder et al. (2019) | L | 70 |
| (methyl iodide) | $2.0\times10^{-3}$ | 3600 | Burkholder et al. (2015) | L | |
| [74-88-4] | $1.5\times10^{-3}$ | 3900 | Burkholder et al. (2015) | L | 70 |
| INQOMBQAUSQDDS-UHFFFAOYSA-N | $2.0\times10^{-3}$ | 3100 | Brockbank (2013) | L | 1 |
| | $2.0\times10^{-3}$ | 3600 | Sander et al. (2011) | L | |
| | $2.0\times10^{-3}$ | 3600 | Sander et al. (2006) | L | |
| | $2.0\times10^{-3}$ | 3600 | Staudinger and Roberts (2001) | L | |
| | $1.8\times10^{-3}$ | 3200 | Hiatt (2013) | M | |
| | $1.4\times10^{-3}$ | 3900 | Ooki and Yokouchi (2011) | M | 70 |
| | $1.9\times10^{-3}$ | | Gan and Yates (1996) | M | 294 |
| | $1.4\times10^{-3}$ | 4100 | Moore et al. (1995) | M | 794, 70 |
| | $2.0\times10^{-3}$ | 3700 | Elliott and Rowland (1993) | M | |
| | $1.9\times10^{-3}$ | 3800 | Hunter-Smith et al. (1983) | M | 658 |
| | $2.0\times10^{-3}$ | 3100 | Balls (1980) | M | |
| | $1.8\times10^{-3}$ | 3000 | Swain and Thornton (1962) | M | |
| | $1.9\times10^{-3}$ | 3000 | Glew and Moelwyn-Hughes (1953) | M | 795 |
| | $1.9\times10^{-3}$ | 3700 | Rex (1906) | M | |
| | $1.8\times10^{-3}$ | | Mackay et al. (2006b) | V | |
| | $1.9\times10^{-3}$ | 3600 | Fogg and Sangster (2003) | V | |
| | $1.8\times10^{-3}$ | | Mackay et al. (1993) | V | |
| | $1.8\times10^{-3}$ | | Abraham (1984) | V | |
| | $1.8\times10^{-3}$ | | Hine and Mookerjee (1975) | V | |
| | $3.5\times10^{-3}$ | | Yaws (2003) | X | 237 |
| | $1.7\times10^{-3}$ | | Liss and Slater (1974) | C | |
| | $8.8\times10^{-3}$ | | Keshavarz et al. (2022) | Q | |
| | $3.7\times10^{-3}$ | | Duchowicz et al. (2020) | Q | |
| | $7.4\times10^{-3}$ | | Gharagheizi et al. (2012) | Q | |
| | $1.6\times10^{-3}$ | | Raventos-Duran et al. (2010) | Q | 271, 243 |
| | $1.2\times10^{-3}$ | | Raventos-Duran et al. (2010) | Q | 244 |
| | $2.0\times10^{-3}$ | | Raventos-Duran et al. (2010) | Q | 245 |
| | $3.4\times10^{-3}$ | | Gharagheizi et al. (2010) | Q | 246 |
| | $2.1\times10^{-3}$ | | Hilal et al. (2008) | Q | |
| | $6.8\times10^{-4}$ | | Modarresi et al. (2007) | Q | 67 |
| | | 3800 | Kühne et al. (2005) | Q | |
| | $1.9\times10^{-3}$ | | Yaffe et al. (2003) | Q | 248, 249 |
| | $2.1\times10^{-4}$ | | English and Carroll (2001) | Q | 230, 231 |
| | $8.8\times10^{-4}$ | | Katritzky et al. (1998) | Q | |
| | $1.8\times10^{-3}$ | | Suzuki et al. (1992) | Q | 232 |
| | $3.6\times10^{-3}$ | | Nirmalakhandan and Speece (1988) | Q | |
| | $1.9\times10^{-3}$ | | Duchowicz et al. (2020) | ? | 185, 21 |
| | $1.8\times10^{-3}$ | | Mackay et al. (2006b) | ? | |
| | | 3700 | Kühne et al. (2005) | ? | |





Table A8.1: Iodocarbons (C, H, O, Cl, I) (. . . continued)

| Substance Formula (Trivial Name) [CAS Registry Number] InChIKey | $H_s^{cp}$ (at $T^\ominus$) $\left[\dfrac{\mathrm{mol}}{\mathrm{m^3\,Pa}}\right]$ | $\dfrac{\mathrm{d}\ln H_s^{cp}}{\mathrm{d}(1/T)}$ [K] | Reference | Type | Note |
|---|---|---|---|---|---|
| | $3.5\times10^{-3}$ | | Yaws (1999) | ? | 21 |
| | $1.8\times10^{-3}$ | | Mackay et al. (1993) | ? | |
| | $3.5\times10^{-3}$ | | Yaws and Yang (1992) | ? | 21 |
| diiodomethane | $2.3\times10^{-2}$ | 5300 | Burkholder et al. (2019) | L | 70 |
| $CH_2I_2$ | $2.3\times10^{-2}$ | 5300 | Burkholder et al. (2015) | L | 70 |
| [75-11-6] | $2.4\times10^{-2}$ | 4700 | Moore et al. (1995) | M | 796, 70 |
| NZZFYRREKKOMAT-UHFFFAOYSA-N | $3.2\times10^{-2}$ | | Mackay et al. (1993) | V | |
| | $9.9\times10^{-3}$ | | Raventos-Duran et al. (2010) | Q | 271, 243 |
| | $6.2\times10^{-2}$ | | Raventos-Duran et al. (2010) | Q | 244 |
| | $2.5\times10^{-2}$ | | Raventos-Duran et al. (2010) | Q | 245 |
| | $7.3\times10^{-2}$ | | Hilal et al. (2008) | Q | |
| | $1.3\times10^{-3}$ | | Modarresi et al. (2007) | Q | 67 |
| | $2.7\times10^{-2}$ | | Yao et al. (2002) | Q | 229 |
| | $3.1\times10^{-2}$ | | Katritzky et al. (1998) | Q | |
| | $2.9\times10^{-2}$ | | Yaws (1999) | ? | 21 |
| | $2.8\times10^{-2}$ | | Yaws and Yang (1992) | ? | 21 |
| | $2.8\times10^{-2}$ | | Abraham et al. (1990) | ? | |
| triiodomethane | $6.2\times10^{-3}$ | | Fogg and Sangster (2003) | V | |
| $CHI_3$ | $4.3\times10^{-3}$ | | Yaws (2003) | X | 237 |
| (iodoform) | $3.2\times10^{-6}$ | | HSDB (2015) | Q | 99 |
| [75-47-8] | $4.3\times10^{-3}$ | | Gharagheizi et al. (2010) | Q | 246 |
| OKJPEAGHQZHRQV-UHFFFAOYSA-N | $1.3\times10^{-3}$ | | Hilal et al. (2008) | Q | |
| | $3.4\times10^{-3}$ | | Yaws and Yang (1992) | ? | 21 |
| iodoethane | $1.4\times10^{-3}$ | 4800 | Burkholder et al. (2019) | L | 70 |
| $C_2H_5I$ | $1.4\times10^{-3}$ | 4800 | Burkholder et al. (2015) | L | 70 |
| [75-03-6] | $1.5\times10^{-3}$ | 3600 | Brockbank (2013) | L | 1 |
| HVTICUPFWKNHNG-UHFFFAOYSA-N | $1.5\times10^{-3}$ | 4200 | Fogg and Sangster (2003) | L | 797 |
| | $1.4\times10^{-3}$ | 4800 | Ooki and Yokouchi (2011) | M | 70 |
| | $1.4\times10^{-3}$ | | Li et al. (1993) | M | |
| | $1.5\times10^{-3}$ | 4000 | Rex (1906) | M | |
| | $1.4\times10^{-3}$ | | Duchowicz et al. (2020) | V | 186 |
| | $1.4\times10^{-3}$ | | Mackay et al. (2006b) | V | |
| | $1.9\times10^{-3}$ | | Mackay et al. (1993) | V | |
| | $1.4\times10^{-3}$ | | Abraham (1984) | V | |
| | $1.4\times10^{-3}$ | | Hine and Mookerjee (1975) | V | |
| | $1.7\times10^{-3}$ | | Yaws (2003) | X | 237, 12 |
| | $1.2\times10^{-3}$ | | Duchowicz et al. (2020) | Q | |
| | $1.2\times10^{-3}$ | | Raventos-Duran et al. (2010) | Q | 242, 243 |
| | $1.2\times10^{-3}$ | | Raventos-Duran et al. (2010) | Q | 244 |
| | $1.2\times10^{-3}$ | | Raventos-Duran et al. (2010) | Q | 245 |
| | $1.7\times10^{-3}$ | | Gharagheizi et al. (2010) | Q | 246 |
| | $1.9\times10^{-3}$ | | Hilal et al. (2008) | Q | |
| | | 4200 | Kühne et al. (2005) | Q | |
| | $1.1\times10^{-3}$ | | Yaffe et al. (2003) | Q | 248, 272 |
| | $4.2\times10^{-3}$ | | Yao et al. (2002) | Q | 229, 267 |
| | $2.2\times10^{-4}$ | | English and Carroll (2001) | Q | 230, 231 |



Table A8.1: Iodocarbons (C, H, O, Cl, I) (...continued)

| Substance<br>Formula<br>(Trivial Name)<br>[CAS Registry Number]<br>InChIKey | $H_s^{cp}$<br>(at $T^\ominus$)<br>$\left[\dfrac{\mathrm{mol}}{\mathrm{m}^3\,\mathrm{Pa}}\right]$ | $\dfrac{\mathrm{d}\ln H_s^{cp}}{\mathrm{d}(1/T)}$<br><br>[K] | Reference | Type | Note |
|---|---|---|---|---|---|
| | $1.2\times10^{-3}$ | | Katritzky et al. (1998) | Q | |
| | $1.3\times10^{-3}$ | | Suzuki et al. (1992) | Q | 232 |
| | $1.2\times10^{-3}$ | | Nirmalakhandan and Speece (1988) | Q | |
| | | 4100 | Kühne et al. (2005) | ? | |
| | $1.4\times10^{-3}$ | | Yaws (1999) | ? | 21, 12 |
| | $1.1\times10^{-3}$ | | Abraham and Weathersby (1994) | ? | 21 |
| | $1.8\times10^{-3}$ | | Yaws and Yang (1992) | ? | 21, 12 |
| | $1.4\times10^{-3}$ | | Abraham et al. (1990) | ? | |
| (*E*)-1,2-diiodoethene<br>$C_2H_2I_2$<br>[590-27-2]<br>CVOGMKGEVNGRSK-OWOJBTEDSA-N | $3.5\times10^{-3}$<br>$5.9\times10^{-3}$ | | Duchowicz et al. (2020)<br>Duchowicz et al. (2020) | V<br>Q | 186 |
| 1-iodopropane<br>$C_3H_7I$<br>(propyl iodide)<br>[107-08-4]<br>PVWOIHVRPOBWPI-UHFFFAOYSA-N | $1.1\times10^{-3}$ | 4500 | Brockbank (2013) | L | 1 |
| | $1.1\times10^{-3}$ | | Li et al. (1993) | M | |
| | $1.0\times10^{-3}$ | 4600 | Rex (1906) | M | |
| | $1.1\times10^{-3}$ | | Duchowicz et al. (2020) | V | 186 |
| | $1.1\times10^{-3}$ | | Mackay et al. (2006b) | V | |
| | $1.1\times10^{-3}$ | | Mackay et al. (1993) | V | |
| | $9.9\times10^{-4}$ | | Abraham (1984) | V | |
| | $1.1\times10^{-3}$ | | Hine and Mookerjee (1975) | V | |
| | $1.2\times10^{-3}$ | | Yaws (2003) | X | 237, 79 |
| | $1.2\times10^{-3}$ | | Duchowicz et al. (2020) | Q | |
| | $2.9\times10^{-3}$ | | Gharagheizi et al. (2012) | Q | |
| | $9.9\times10^{-4}$ | | Raventos-Duran et al. (2010) | Q | 271, 243 |
| | $9.9\times10^{-4}$ | | Raventos-Duran et al. (2010) | Q | 244 |
| | $9.9\times10^{-4}$ | | Raventos-Duran et al. (2010) | Q | 245 |
| | $1.0\times10^{-3}$ | | Gharagheizi et al. (2010) | Q | 246 |
| | $1.6\times10^{-3}$ | | Hilal et al. (2008) | Q | |
| | $2.4\times10^{-4}$ | | Modarresi et al. (2007) | Q | 67 |
| | | 4500 | Kühne et al. (2005) | Q | |
| | $1.1\times10^{-3}$ | | Yaffe et al. (2003) | Q | 248, 249 |
| | $1.9\times10^{-3}$ | | Yao et al. (2002) | Q | 229 |
| | $1.1\times10^{-3}$ | | English and Carroll (2001) | Q | 230, 231 |
| | $1.1\times10^{-3}$ | | Katritzky et al. (1998) | Q | |
| | $9.9\times10^{-4}$ | | Suzuki et al. (1992) | Q | 232 |
| | $9.5\times10^{-4}$ | | Nirmalakhandan and Speece (1988) | Q | |
| | | 4500 | Kühne et al. (2005) | ? | |
| | $1.1\times10^{-3}$ | | Yaws (1999) | ? | 21, 79 |
| | $1.2\times10^{-3}$ | | Yaws and Yang (1992) | ? | 21, 79 |
| | $9.9\times10^{-4}$ | | Abraham et al. (1990) | ? | |





Table A8.1: Iodocarbons (C, H, O, Cl, I) (. . . continued)

| Substance<br>Formula<br>(Trivial Name)<br>[CAS Registry Number]<br>InChIKey | $H_s^{cp}$<br>(at $T^{\ominus}$)<br>$\left[\dfrac{\mathrm{mol}}{\mathrm{m^3\,Pa}}\right]$ | $\dfrac{\mathrm{d}\ln H_s^{cp}}{\mathrm{d}(1/T)}$<br><br>[K] | Reference | Type | Note |
|---|---|---|---|---|---|
| 2-iodopropane | $8.5\times10^{-4}$ | 4700 | Brockbank (2013) | L | 1 |
| $C_3H_7I$ | $8.5\times10^{-4}$ | 4500 | Rex (1906) | M | |
| (isopropyl iodide) | $1.4\times10^{-3}$ | | Duchowicz et al. (2020) | V | 186 |
| [75-30-9] | $8.8\times10^{-4}$ | | Hine and Mookerjee (1975) | V | |
| FMKOJHQASLBPH-UHFFFAOYSA-N | $1.1\times10^{-3}$ | | Yaws (2003) | X | 237, 12 |
| | $5.7\times10^{-4}$ | | Duchowicz et al. (2020) | Q | |
| | $6.8\times10^{-3}$ | | Gharagheizi et al. (2012) | Q | |
| | $9.9\times10^{-4}$ | | Raventos-Duran et al. (2010) | Q | 242, 243 |
| | $4.9\times10^{-4}$ | | Raventos-Duran et al. (2010) | Q | 244 |
| | $9.9\times10^{-4}$ | | Raventos-Duran et al. (2010) | Q | 245 |
| | $1.1\times10^{-3}$ | | Gharagheizi et al. (2010) | Q | 246 |
| | $7.9\times10^{-4}$ | | Hilal et al. (2008) | Q | |
| | $1.7\times10^{-4}$ | | Modarresi et al. (2007) | Q | 67 |
| | | 4500 | Kühne et al. (2005) | Q | |
| | $1.5\times10^{-3}$ | | Yaffe et al. (2003) | Q | 248, 249 |
| | $1.8\times10^{-3}$ | | Yao et al. (2002) | Q | 229 |
| | $7.9\times10^{-4}$ | | English and Carroll (2001) | Q | 230, 231 |
| | $9.7\times10^{-4}$ | | Katritzky et al. (1998) | Q | |
| | $8.6\times10^{-4}$ | | Suzuki et al. (1992) | Q | 232 |
| | $5.4\times10^{-4}$ | | Nirmalakhandan and Speece (1988) | Q | |
| | | 4700 | Kühne et al. (2005) | ? | |
| | $8.7\times10^{-4}$ | | Yaws (1999) | ? | 21, 12 |
| | $1.1\times10^{-3}$ | | Yaws and Yang (1992) | ? | 21, 12 |
| 1-iodobutane | $6.9\times10^{-4}$ | | Brockbank (2013) | L | |
| $C_4H_9I$ | $5.9\times10^{-4}$ | | Duchowicz et al. (2020) | V | 186 |
| [542-69-8] | $5.4\times10^{-4}$ | | Mackay et al. (2006b) | V | |
| KMGBZBJJOKUPIA-UHFFFAOYSA-N | $5.4\times10^{-4}$ | | Mackay et al. (1993) | V | |
| | $6.1\times10^{-4}$ | | Abraham (1984) | V | |
| | $6.2\times10^{-4}$ | | Hine and Mookerjee (1975) | V | |
| | $1.2\times10^{-3}$ | | Duchowicz et al. (2020) | Q | |
| | $6.2\times10^{-4}$ | | Raventos-Duran et al. (2010) | Q | 242, 243 |
| | $6.2\times10^{-4}$ | | Raventos-Duran et al. (2010) | Q | 244 |
| | $7.8\times10^{-4}$ | | Raventos-Duran et al. (2010) | Q | 245 |
| | $1.2\times10^{-3}$ | | Hilal et al. (2008) | Q | |
| | $1.8\times10^{-4}$ | | Modarresi et al. (2007) | Q | 67 |
| | $6.7\times10^{-4}$ | | Yaffe et al. (2003) | Q | 248, 249 |
| | $7.9\times10^{-4}$ | | English and Carroll (2001) | Q | 230, 231 |
| | $1.1\times10^{-3}$ | | Katritzky et al. (1998) | Q | |
| | $7.5\times10^{-4}$ | | Suzuki et al. (1992) | Q | 232 |
| | $7.5\times10^{-4}$ | | Nirmalakhandan and Speece (1988) | Q | |
| | $6.1\times10^{-4}$ | | Abraham et al. (1990) | ? | |



Table A8.1: Iodocarbons (C, H, O, Cl, I) (...continued)

| Substance Formula (Trivial Name) [CAS Registry Number] InChIKey | $H_s^{cp}$ (at $T^{\ominus}$) $\left[\dfrac{\text{mol}}{\text{m}^3\,\text{Pa}}\right]$ | $\dfrac{\text{d}\ln H_s^{cp}}{\text{d}(1/T)}$ [K] | Reference | Type | Note |
|---|---|---|---|---|---|
| 2-iodobutane | $5.0\times10^{-4}$ | | Duchowicz et al. (2020) | V | 186 |
| $C_4H_9I$ | $5.7\times10^{-4}$ | | Duchowicz et al. (2020) | Q | |
| [513-48-4] | $6.2\times10^{-4}$ | | Raventos-Duran et al. (2010) | Q | 242, 243 |
| IQRUSQUYPCHEKN-UHFFFAOYSA-N | $4.9\times10^{-4}$ | | Raventos-Duran et al. (2010) | Q | 244 |
| | $7.8\times10^{-4}$ | | Raventos-Duran et al. (2010) | Q | 245 |
| | $7.0\times10^{-4}$ | | Hilal et al. (2008) | Q | |
| 1-iodopentane | $5.9\times10^{-4}$ | | Keshavarz et al. (2022) | Q | |
| $C_5H_{11}I$ | $1.2\times10^{-3}$ | | Duchowicz et al. (2020) | Q | 184 |
| [628-17-1] | $4.9\times10^{-4}$ | | Raventos-Duran et al. (2010) | Q | 242, 243 |
| BLXSFCHWMBESKV-UHFFFAOYSA-N | $4.9\times10^{-4}$ | | Raventos-Duran et al. (2010) | Q | 244 |
| | $6.2\times10^{-4}$ | | Raventos-Duran et al. (2010) | Q | 245 |
| | $9.9\times10^{-4}$ | | Hilal et al. (2008) | Q | |
| | $1.4\times10^{-4}$ | | Modarresi et al. (2007) | Q | 67 |
| | $5.8\times10^{-4}$ | | English and Carroll (2001) | Q | 230, 231 |
| | $5.7\times10^{-4}$ | | Nirmalakhandan et al. (1997) | Q | |
| | $5.1\times10^{-4}$ | | Duchowicz et al. (2020) | ? | 185, 21 |
| | $5.1\times10^{-4}$ | | Abraham et al. (1990) | ? | |
| 1-iodohexane | $3.4\times10^{-4}$ | | Brockbank (2013) | L | |
| $C_6H_{13}I$ | $8.0\times10^{-4}$ | | Keshavarz et al. (2022) | Q | |
| [638-45-9] | $1.2\times10^{-3}$ | | Duchowicz et al. (2020) | Q | |
| ANOOTOPTCJRUPK-UHFFFAOYSA-N | $8.2\times10^{-4}$ | | Hilal et al. (2008) | Q | |
| | $1.2\times10^{-4}$ | | Modarresi et al. (2007) | Q | 67 |
| | $4.2\times10^{-4}$ | | English and Carroll (2001) | Q | 230, 260 |
| | $4.5\times10^{-4}$ | | Nirmalakhandan et al. (1997) | Q | |
| | $3.5\times10^{-4}$ | | Duchowicz et al. (2020) | ? | 185, 21 |
| | $3.5\times10^{-4}$ | | Abraham et al. (1990) | ? | |
| 1-iodoheptane | $2.6\times10^{-4}$ | | Abraham (1984) | V | |
| $C_7H_{15}I$ | $1.1\times10^{-3}$ | | Keshavarz et al. (2022) | Q | |
| [4282-40-0] | $1.2\times10^{-3}$ | | Duchowicz et al. (2020) | Q | 184 |
| LMHCYRULPLGEEZ-UHFFFAOYSA-N | $3.1\times10^{-4}$ | | Raventos-Duran et al. (2010) | Q | 242, 243 |
| | $3.1\times10^{-4}$ | | Raventos-Duran et al. (2010) | Q | 244 |
| | $3.1\times10^{-4}$ | | Raventos-Duran et al. (2010) | Q | 245 |
| | $6.7\times10^{-4}$ | | Hilal et al. (2008) | Q | |
| | $9.7\times10^{-5}$ | | Modarresi et al. (2007) | Q | 67 |
| | $3.1\times10^{-4}$ | | English and Carroll (2001) | Q | 230, 231 |
| | $3.5\times10^{-4}$ | | Nirmalakhandan et al. (1997) | Q | |
| | $2.5\times10^{-4}$ | | Duchowicz et al. (2020) | ? | 185, 21 |
| | $2.5\times10^{-4}$ | | Abraham et al. (1990) | ? | |
| iodocyclohexane | $3.9\times10^{-3}$ | | Hilal et al. (2008) | Q | |
| $C_6H_{11}I$ | | | | | |
| [626-62-0] | | | | | |
| FUCOMWZKWIEKRK-UHFFFAOYSA-N | | | | | |



Table A8.1: Iodocarbons (C, H, O, Cl, I) (... continued)

| Substance Formula (Trivial Name) [CAS Registry Number] InChIKey | $H_s^{cp}$ (at $T^\ominus$) $\left[\dfrac{\text{mol}}{\text{m}^3\,\text{Pa}}\right]$ | $\dfrac{\text{d}\ln H_s^{cp}}{\text{d}(1/T)}$ [K] | Reference | Type | Note |
|---|---|---|---|---|---|
| 3-iodo-1-propene<br>$C_3H_5I$<br>[556-56-9]<br>HFEHLDPGIKPNKL-UHFFFAOYSA-N | $3.8\times10^{-3}$ | | Hilal et al. (2008) | Q | |
| 1-iodocyclohexene<br>$C_6H_9I$<br>[17497-53-9]<br>CAROGICRCNKEOD-UHFFFAOYSA-N | $4.1\times10^{-3}$ | | Hilal et al. (2008) | Q | |
| iodobenzene<br>$C_6H_5I$<br>[591-50-4]<br>SNHMUERNLJLMHN-UHFFFAOYSA-N | $7.7\times10^{-3}$<br>$7.6\times10^{-3}$<br>$1.2\times10^{-2}$<br>$1.2\times10^{-2}$<br>$7.9\times10^{-3}$<br>$1.3\times10^{-2}$<br>$6.9\times10^{-3}$<br>$8.6\times10^{-3}$<br>$7.7\times10^{-3}$<br>$8.1\times10^{-2}$<br>$7.8\times10^{-3}$<br>$2.0\times10^{-2}$<br>$7.8\times10^{-3}$<br>$6.9\times10^{-3}$<br>$1.4\times10^{-2}$<br>$6.9\times10^{-3}$<br>$7.7\times10^{-3}$<br>$8.3\times10^{-3}$<br>$2.3\times10^{-3}$<br>$6.4\times10^{-3}$<br>$3.8\times10^{-3}$<br>$7.0\times10^{-3}$<br>$7.4\times10^{-3}$<br>$7.7\times10^{-3}$ | | Mackay and Shiu (1981)<br>Li and Carr (1993)<br>Duchowicz et al. (2020)<br>HSDB (2015)<br>Schüürmann (2000)<br>Mackay et al. (1993)<br>Yaws (2003)<br>Duchowicz et al. (2020)<br>Li et al. (2014)<br>Gharagheizi et al. (2012)<br>Raventos-Duran et al. (2010)<br>Raventos-Duran et al. (2010)<br>Raventos-Duran et al. (2010)<br>Gharagheizi et al. (2010)<br>Hilal et al. (2008)<br>Modarresi et al. (2007)<br>Yaffe et al. (2003)<br>Yao et al. (2002)<br>English and Carroll (2001)<br>Katritzky et al. (1998)<br>Nirmalakhandan et al. (1997)<br>Yaws (1999)<br>Yaws and Yang (1992)<br>Abraham et al. (1990)<br>Brockbank (2013) | L<br>M<br>V<br>V<br>V<br>V<br>X<br>Q<br>Q<br>Q<br>Q<br>Q<br>Q<br>Q<br>Q<br>Q<br>Q<br>Q<br>Q<br>Q<br>Q<br>?<br>?<br>?<br>W | <br><br>186<br><br><br><br>237<br><br>241<br><br>242, 243<br>244<br>245<br>246<br><br>67<br>248, 249<br>229<br>230, 231<br><br><br>21<br>21<br><br>798 |
| iodoacetic acid<br>$C_2H_3IO_2$<br>[64-69-7]<br>JDNTWHVOXJZDSN-UHFFFAOYSA-N | $2.4\times10^2$ | | HSDB (2015) | Q | 99 |
| 2-iodophenol<br>$C_6H_5IO$<br>[533-58-4]<br>KQDJTBPASNJQFQ-UHFFFAOYSA-N | $1.4\times10^1$<br>6.9<br>$1.4\times10^1$<br>$1.5\times10^1$<br>$1.6\times10^2$ | | Abraham et al. (1994a)<br>Hilal et al. (2008)<br>Yaffe et al. (2003)<br>English and Carroll (2001)<br>Nirmalakhandan et al. (1997) | R<br>Q<br>Q<br>Q<br>Q | <br><br>248, 249<br>230, 231<br> |



Table A8.1: Iodocarbons (C, H, O, Cl, I) (...continued)

| Substance<br>Formula<br>(Trivial Name)<br>[CAS Registry Number]<br>InChIKey | $H_s^{cp}$<br>(at $T^\ominus$)<br>$\left[\dfrac{\text{mol}}{\text{m}^3\,\text{Pa}}\right]$ | $\dfrac{\text{d}\ln H_s^{cp}}{\text{d}(1/T)}$<br><br>[K] | Reference | Type | Note |
|---|---|---|---|---|---|
| 3-iodophenol<br>$C_6H_5IO$<br>[626-02-8]<br>FXTKWBZFNQHAAO-UHFFFAOYSA-N | $7.0\times10^1$ | | Hilal et al. (2008) | Q | |
| 4-iodophenol<br>$C_6H_5IO$<br>[540-38-5]<br>VSMDINRNYYEDRN-UHFFFAOYSA-N | $4.6\times10^1$ | | Hilal et al. (2008) | Q | |
| erythrosine<br>$C_{20}H_8I_4O_5$<br>[16423-68-0]<br>OALHHIHQOFIMEF-UHFFFAOYSA-N | $3.9\times10^{13}$<br>$2.3\times10^8$<br>$8.6\times10^{10}$<br>$5.1\times10^9$ | | Zhang et al. (2010)<br>Zhang et al. (2010)<br>Zhang et al. (2010)<br>Zhang et al. (2010) | Q<br>Q<br>Q<br>Q | 287, 288<br>287, 289<br>287, 290<br>287, 291 |
| 4-hydroxy-3,5-diiodo-benzonitrile<br>$C_7H_3I_2NO$<br>(ioxynil)<br>[1689-83-4]<br>NRXQIUSYPAHGNM-UHFFFAOYSA-N | $1.3\times10^2$<br>$1.8\times10^4$ | | Mackay et al. (2006d)<br>HSDB (2015) | V<br>Q | <br>99 |
| 3-iodo-2-propynyl butylcarbamate<br>$C_8H_{12}INO_2$<br>[55406-53-6]<br>WYVVKGNFXHOCQV-UHFFFAOYSA-N | $8.2\times10^1$ | | HSDB (2015) | V | |
| diatrizoic acid<br>$C_{11}H_9I_3N_2O_4$<br>[117-96-4]<br>YVPYQUNUQOZFHG-UHFFFAOYSA-N | $3.5\times10^{12}$<br>$5.4\times10^8$<br>$1.2\times10^{17}$<br>$3.3\times10^{16}$ | | Zhang et al. (2010)<br>Zhang et al. (2010)<br>Zhang et al. (2010)<br>Zhang et al. (2010) | Q<br>Q<br>Q<br>Q | 287, 288<br>287, 289<br>287, 290<br>287, 291 |
| iothalamic acid<br>$C_{11}H_9I_3N_2O_4$<br>[2276-90-6]<br>UXIGWFXRQKWHHA-UHFFFAOYSA-N | $4.4\times10^{12}$<br>$4.8\times10^9$<br>$4.2\times10^{16}$<br>$1.9\times10^{16}$ | | Zhang et al. (2010)<br>Zhang et al. (2010)<br>Zhang et al. (2010)<br>Zhang et al. (2010) | Q<br>Q<br>Q<br>Q | 287, 288<br>287, 289<br>287, 290<br>287, 291 |
| benodanil<br>$C_{13}H_{10}INO$<br>[15310-01-7]<br>LJOZMWRYMKECFF-UHFFFAOYSA-N | $6.2\times10^5$<br>$>2.3\times10^{10}$ | | Mackay et al. (2006d)<br>MacBean (2012a) | V<br>? | |
| proquinazid<br>$C_{14}H_{17}IN_2O_2$<br>[189278-12-4]<br>FLVBXVXXXMLMOX-UHFFFAOYSA-N | $3.3\times10^1$ | | Maniere et al. (2011) | ? | 165 |
| iopamidol<br>$C_{17}H_{22}I_3N_3O_8$<br>[60166-93-0]<br>XQZXYNRDCRIARQ-LURJTMIESA-N | $9.0\times10^{19}$ | | HSDB (2015) | Q | 99 |



Table A8.1: Iodocarbons (C, H, O, Cl, I) (...continued)

| Substance<br>Formula<br>(Trivial Name)<br>[CAS Registry Number]<br>InChIKey | $H_s^{cp}$<br>(at $T^\ominus$)<br>$\left[\dfrac{\mathrm{mol}}{\mathrm{m^3\,Pa}}\right]$ | $\dfrac{\mathrm{d}\ln H_s^{cp}}{\mathrm{d}(1/T)}$<br><br>[K] | Reference | Type | Note |
|---|---|---|---|---|---|
| ioxaglic acid<br>$C_{24}H_{21}I_6N_5O_8$<br>[59017-64-0]<br>TYYBFXNZMFNZJT-UHFFFAOYSA-N | $2.7\times10^{35}$<br>$1.4\times10^{27}$<br>$2.0\times10^{29}$<br>$7.2\times10^{38}$ | | Zhang et al. (2010)<br>Zhang et al. (2010)<br>Zhang et al. (2010)<br>Zhang et al. (2010) | Q<br>Q<br>Q<br>Q | 287, 288<br>287, 289<br>287, 290<br>287, 291 |
| perfluorohexyl iodide<br>$C_6F_{13}I$<br>[355-43-1]<br>BULLJMKUVKYZDJ-UHFFFAOYSA-N | $4.6\times10^{-4}$ | 8200 | Abusallout et al. (2022) | M | |
| 1,1,1,2,2,3,3-heptafluoro-5-<br>iodopentane<br>$C_5H_4F_7I$<br>[68188-12-5]<br>TZNRRNKRZXHADL-UHFFFAOYSA-N | $4.6\times10^{-6}$<br><br>$1.2\times10^{-4}$<br>$3.8\times10^{-4}$<br>$5.0\times10^{-6}$ | | Zhang et al. (2010)<br><br>Zhang et al. (2010)<br>Zhang et al. (2010)<br>Zhang et al. (2010) | Q<br><br>Q<br>Q<br>Q | 287, 288<br><br>287, 289<br>287, 290<br>287, 291 |
| 1-fluoro-4-iodobenzene<br>$C_6H_4FI$<br>[352-34-1]<br>KGNQDBQYEBMPFZ-UHFFFAOYSA-N | $4.5\times10^{-3}$ | | Ebert et al. (2023) | ? | 316 |
| 5-diethylamiloride<br>$C_6H_4F_9I$<br>[2043-55-2]<br>CXHFIVFPHDGZIS-UHFFFAOYSA-N | $8.8\times10^{-7}$<br>$5.6\times10^{-5}$<br>$1.9\times10^{-4}$<br>$1.0\times10^{-6}$ | | Zhang et al. (2010)<br>Zhang et al. (2010)<br>Zhang et al. (2010)<br>Zhang et al. (2010) | Q<br>Q<br>Q<br>Q | 287, 288<br>287, 289<br>287, 290<br>287, 291 |
| 1,1,1,2,2,3,3,4,4,5,5,6,6-<br>tridecafluoro-8-iodooctane<br>$C_8H_4F_{13}I$<br>[2043-57-4]<br>NVVZEKTVIXIUKW-UHFFFAOYSA-N | $1.3\times10^{-3}$<br><br>$3.2\times10^{-8}$<br>$3.4\times10^{-6}$<br>$5.4\times10^{-5}$<br>$4.3\times10^{-8}$ | | Abusallout et al. (2022)<br><br>Zhang et al. (2010)<br>Zhang et al. (2010)<br>Zhang et al. (2010)<br>Zhang et al. (2010) | M<br><br>Q<br>Q<br>Q<br>Q | <br><br>287, 288<br>287, 289<br>287, 290<br>287, 291 |
| 1H,2H-perfluoro-1-iodooct-1-ene<br>$C_8H_2F_{13}I$<br>[150223-14-6]<br>WDWNMIBWWPZNJK-OWOJBTEDSA-N | $8.4\times10^{-4}$ | | Abusallout et al. (2022) | M | |
| 1,1,1,2,2,3,3,4,4,5,5,6,6,7,7,8,8-<br>heptadecafluoro-10-iododecane<br>$C_{10}H_4F_{17}I$<br>[2043-53-0]<br>XVKJSLBVVRCOIT-UHFFFAOYSA-N | $1.2\times10^{-9}$<br><br>$7.7\times10^{-8}$<br>$2.0\times10^{-5}$<br>$2.3\times10^{-9}$ | | Zhang et al. (2010)<br><br>Zhang et al. (2010)<br>Zhang et al. (2010)<br>Zhang et al. (2010) | Q<br><br>Q<br>Q<br>Q | 287, 288<br><br>287, 289<br>287, 290<br>287, 291 |
| chloroiodomethane<br>$CH_2ClI$<br>[593-71-5]<br>PJGJQVRXEUVAFT-UHFFFAOYSA-N | $8.4\times10^{-3}$<br>$8.4\times10^{-3}$<br>$8.3\times10^{-3}$<br>$9.1\times10^{-3}$<br>$4.9\times10^{-3}$<br>$2.5\times10^{-2}$<br>$4.9\times10^{-3}$ | 5100<br>5100<br>6200<br>4100 | Burkholder et al. (2019)<br>Burkholder et al. (2015)<br>Ooki and Yokouchi (2011)<br>Moore et al. (1995)<br>Raventos-Duran et al. (2010)<br>Raventos-Duran et al. (2010)<br>Raventos-Duran et al. (2010) | L<br>L<br>M<br>M<br>Q<br>Q<br>Q | 70<br>70<br>70<br>799, 70<br>271, 243<br>244<br>245 |





Table A8.1: Iodocarbons (C, H, O, Cl, I) (...continued)

| Substance Formula (Trivial Name) [CAS Registry Number] InChIKey | $H_s^{cp}$ (at $T^\ominus$) $\left[\dfrac{\text{mol}}{\text{m}^3\,\text{Pa}}\right]$ | $\dfrac{\text{d}\ln H_s^{cp}}{\text{d}(1/T)}$ [K] | Reference | Type | Note |
|---|---|---|---|---|---|
| | $2.0\times10^{-2}$ | | Hilal et al. (2008) | Q | |
| | $5.0\times10^{-4}$ | | Modarresi et al. (2007) | Q | 67 |
| 1-chloro-4-iodobenzene $C_6H_4ClI$ [637-87-6] GWQSENYKCGJTRI-UHFFFAOYSA-N | $6.9\times10^{-3}$ | | Ebert et al. (2023) | ? | 316 |
| bromoiodomethane $CH_2BrI$ [557-68-6] TUDWMIUPYRKEFN-UHFFFAOYSA-N | $2.0\times10^{-2}$ | | Karagodin-Doyennel et al. (2021) | E | 800 |
| 1-bromo-4-iodobenzene $C_6H_4BrI$ [589-87-7] UCCUXODGPMAHRL-UHFFFAOYSA-N | $1.5\times10^{-2}$ | | Ebert et al. (2023) | ? | 316 |





## A9 Organic species with sulfur (S)

### A9.1 Sulfur (C, H, O, N, Cl, S)

Table A9.1: Sulfur (C, H, O, N, Cl, S)

| Substance Formula (Trivial Name) [CAS Registry Number] InChIKey | $H_s^{cp}$ (at $T^{\ominus}$) $\left[\dfrac{\text{mol}}{\text{m}^3\,\text{Pa}}\right]$ | $\dfrac{\text{d}\ln H_s^{cp}}{\text{d}(1/T)}$ [K] | Reference | Type | Note |
|---|---|---|---|---|---|
| methanethiol | $3.8\times10^{-3}$ | 3400 | Burkholder et al. (2019) | L | |
| $CH_3SH$ | $3.8\times10^{-3}$ | 3400 | Burkholder et al. (2015) | L | |
| (methyl mercaptan) | $3.8\times10^{-3}$ | 3400 | Sander et al. (2011) | L | |
| [74-93-1] | $3.8\times10^{-3}$ | 3400 | Sander et al. (2006) | L | |
| LSDPWZHWYPCBBB-UHFFFAOYSA-N | $2.8\times10^{-3}$ | 2900 | Plyasunova et al. (2004) | L | |
| | $2.8\times10^{-3}$ | 3100 | Staudinger and Roberts (2001) | L | |
| | $2.0\times10^{-3}$ | 2800 | De Bruyn et al. (1995b) | M | |
| | $2.3\times10^{-3}$ | 2700 | Tsuji et al. (1990) | M | 62 |
| | $3.9\times10^{-3}$ | 3400 | Przyjazny et al. (1983) | M | |
| | $3.3\times10^{-3}$ | | Hine and Weimar (1965) | M | |
| | $3.2\times10^{-3}$ | | Duchowicz et al. (2020) | V | 186 |
| | $3.2\times10^{-3}$ | | HSDB (2015) | V | |
| | $3.3\times10^{-3}$ | | Hine and Mookerjee (1975) | V | |
| | $2.4\times10^{-3}$ | | Yaws (2003) | X | 237 |
| | $2.6\times10^{-3}$ | 1600 | Goldstein (1982) | X | 298 |
| | $1.4\times10^{-1}$ | | Duchowicz et al. (2020) | Q | |
| | $2.3\times10^{-3}$ | | Gharagheizi et al. (2012) | Q | |
| | $3.0\times10^{-3}$ | | Gharagheizi et al. (2010) | Q | 246 |
| | $3.5\times10^{-3}$ | | Hilal et al. (2008) | Q | |
| | | 3300 | Kühne et al. (2005) | Q | |
| | $4.1\times10^{-3}$ | | Yaffe et al. (2003) | Q | 248, 249 |
| | $1.0\times10^{-2}$ | | Yao et al. (2002) | Q | 229 |
| | $2.1\times10^{-3}$ | | Katritzky et al. (1998) | Q | |
| | $2.9\times10^{-3}$ | | Nirmalakhandan et al. (1997) | Q | |
| | $7.0\times10^{-3}$ | | Russell et al. (1992) | Q | 279 |
| | $4.0\times10^{-3}$ | | Suzuki et al. (1992) | Q | 232 |
| | | 3400 | Kühne et al. (2005) | ? | |
| | $2.4\times10^{-3}$ | | Yaws et al. (2003) | ? | 21 |
| | $5.1\times10^{-3}$ | | Yaws (1999) | ? | 21, 80 |
| | $4.0\times10^{-3}$ | | Abraham et al. (1990) | ? | |
| ethanethiol | $2.8\times10^{-3}$ | 3700 | Burkholder et al. (2019) | L | |
| $C_2H_5SH$ | $2.8\times10^{-3}$ | 3700 | Burkholder et al. (2015) | L | |
| (ethyl mercaptan) | $2.8\times10^{-3}$ | 3700 | Sander et al. (2011) | L | |
| [75-08-1] | $2.8\times10^{-3}$ | 3700 | Sander et al. (2006) | L | |
| DNJIEGIFACGWOD-UHFFFAOYSA-N | $2.3\times10^{-3}$ | 3500 | Plyasunova et al. (2004) | L | |
| | $2.4\times10^{-3}$ | 2800 | Jou et al. (2021) | M | |
| | $2.8\times10^{-3}$ | 3700 | Przyjazny et al. (1983) | M | |
| | $2.2\times10^{-3}$ | | Vitenberg et al. (1975) | M | |
| | $3.4\times10^{-3}$ | | Mackay et al. (2006d) | V | |
| | $3.4\times10^{-3}$ | | Mackay et al. (1995) | V | |
| | $3.4\times10^{-3}$ | | Hwang et al. (1992) | V | |
| | $3.6\times10^{-3}$ | | Hine and Mookerjee (1975) | V | |



Table A9.1: Sulfur (C, H, O, N, Cl, S) (...continued)

| Substance Formula (Trivial Name) [CAS Registry Number] InChIKey | $H_s^{cp}$ (at $T^{\ominus}$) $\left[\dfrac{\mathrm{mol}}{\mathrm{m^3\,Pa}}\right]$ | $\dfrac{\mathrm{d}\ln H_s^{cp}}{\mathrm{d}(1/T)}$ [K] | Reference | Type | Note |
|---|---|---|---|---|---|
| | $3.4\times10^{-3}$ | | Yaws (2003) | X | 237 |
| | $1.8\times10^{-3}$ | | Hayer et al. (2022) | Q | 20 |
| | $1.8\times10^{-3}$ | | Gharagheizi et al. (2012) | Q | |
| | $2.9\times10^{-3}$ | | Gharagheizi et al. (2010) | Q | 246 |
| | $3.9\times10^{-3}$ | | Hilal et al. (2008) | Q | |
| | $5.6\times10^{-3}$ | | Modarresi et al. (2007) | Q | 67 |
| | | 3600 | Kühne et al. (2005) | Q | |
| | $2.2\times10^{-3}$ | | Yaffe et al. (2003) | Q | 248, 249 |
| | $5.8\times10^{-3}$ | | Yao et al. (2002) | Q | 229 |
| | $3.6\times10^{-3}$ | | English and Carroll (2001) | Q | 230, 231 |
| | $2.9\times10^{-3}$ | | Katritzky et al. (1998) | Q | |
| | $1.9\times10^{-3}$ | | Nirmalakhandan et al. (1997) | Q | |
| | $1.0\times10^{-2}$ | | Russell et al. (1992) | Q | 279 |
| | $2.9\times10^{-3}$ | | Suzuki et al. (1992) | Q | 232 |
| | | 3700 | Kühne et al. (2005) | ? | |
| | $3.4\times10^{-3}$ | | Yaws et al. (2003) | ? | 21 |
| | $3.4\times10^{-3}$ | | Yaws (1999) | ? | 21 |
| | $3.4\times10^{-3}$ | | Yaws and Yang (1992) | ? | 21 |
| | $2.8\times10^{-3}$ | | Abraham et al. (1990) | ? | |
| 1,2-ethanedithiol $C_2H_6S_2$ [540-63-6] VYMPLPIFKRHAAC-UHFFFAOYSA-N | $8.2\times10^{-2}$ | | HSDB (2015) | Q | 99 |
| thiirane $C_2H_4S$ (ethylene sulfide) [420-12-2] VOVUARRWDCVURC-UHFFFAOYSA-N | $2.8\times10^{-2}$ | | HSDB (2015) | Q | 99 |
| 1-propanethiol $C_3H_7SH$ (propyl mercaptan) [107-03-9] SUVIGLJNEAMWEG-UHFFFAOYSA-N | $1.7\times10^{-3}$ | 3600 | Plyasunova et al. (2004) | L | |
| | $1.8\times10^{-3}$ | 4100 | Haimi et al. (2006) | M | 801 |
| | $1.7\times10^{-3}$ | 3100 | Coquelet and Richon (2005) | M | |
| | $2.4\times10^{-3}$ | 3900 | Przyjazny et al. (1983) | M | |
| | $9.0\times10^{-4}$ | | Mazza (1980) | M | |
| | $2.4\times10^{-3}$ | | Yaws et al. (2003) | V | 802 |
| | $2.4\times10^{-3}$ | | Yaws (2003) | X | 237 |
| | $1.6\times10^{-2}$ | | Keshavarz et al. (2022) | Q | |
| | $4.7\times10^{-2}$ | | Duchowicz et al. (2020) | Q | 299 |
| | $8.2\times10^{-4}$ | | Gharagheizi et al. (2012) | Q | |
| | $2.3\times10^{-3}$ | | Gharagheizi et al. (2010) | Q | 246 |
| | $3.4\times10^{-3}$ | | Hilal et al. (2008) | Q | |
| | | 4000 | Kühne et al. (2005) | Q | |
| | $3.9\times10^{-3}$ | | Yao et al. (2002) | Q | 229, 267 |
| | $2.9\times10^{-3}$ | | English and Carroll (2001) | Q | 230, 260 |
| | $1.5\times10^{-3}$ | | Nirmalakhandan et al. (1997) | Q | |
| | $2.4\times10^{-3}$ | | Duchowicz et al. (2020) | ? | 185, 21 |
| | | 3800 | Kühne et al. (2005) | ? | |



Table A9.1: Sulfur (C, H, O, N, Cl, S) (. . . continued)

| Substance<br>Formula<br>(Trivial Name)<br>[CAS Registry Number]<br>InChIKey | $H_s^{cp}$<br>(at $T^\ominus$)<br>$\left[\dfrac{\mathrm{mol}}{\mathrm{m^3\,Pa}}\right]$ | $\dfrac{\mathrm{d\ln}H_s^{cp}}{\mathrm{d}(1/T)}$<br><br>[K] | Reference | Type | Note |
|---|---|---|---|---|---|
| | $2.2\times10^{-3}$ | | Yaws (1999) | ? | 21 |
| | $2.4\times10^{-3}$ | | Abraham et al. (1990) | ? | |
| 2-propanethiol | $1.3\times10^{-3}$ | 3700 | Brockbank (2013) | L | 1 |
| $C_3H_8S$ | $1.6\times10^{-3}$ | 4300 | Zin et al. (2016) | M | 803 |
| (isopropyl mercaptan) | $1.3\times10^{-3}$ | 3800 | Haimi et al. (2006) | M | 804 |
| [75-33-2] | $2.2\times10^{-3}$ | | Yaws et al. (2003) | V | 802 |
| KJRCEJOSASVSRA-UHFFFAOYSA-N | $2.2\times10^{-3}$ | | Yaws (2003) | X | 237 |
| | $2.1\times10^{-3}$ | | HSDB (2015) | Q | 99 |
| | $1.8\times10^{-3}$ | | Gharagheizi et al. (2012) | Q | |
| | $2.3\times10^{-3}$ | | Gharagheizi et al. (2010) | Q | 246 |
| | $2.1\times10^{-3}$ | | Hilal et al. (2008) | Q | |
| | $2.9\times10^{-3}$ | | Yao et al. (2002) | Q | 229 |
| | $2.4\times10^{-3}$ | | Yaws (1999) | ? | 21 |
| 1-butanethiol | $1.5\times10^{-3}$ | 4300 | Brockbank (2013) | L | 1 |
| $C_4H_9SH$ | $1.4\times10^{-3}$ | 4400 | Plyasunova et al. (2004) | L | |
| (butyl mercaptan) | $1.5\times10^{-3}$ | 4300 | Haimi et al. (2006) | M | 805 |
| [109-79-5] | $1.5\times10^{-3}$ | 3600 | Coquelet and Richon (2005) | M | |
| WQAQPCDUOCURKW-UHFFFAOYSA-N | $2.2\times10^{-3}$ | 4100 | Przyjazny et al. (1983) | M | |
| | $1.1\times10^{-3}$ | | Mackay et al. (2006d) | V | |
| | $1.1\times10^{-3}$ | | Mackay et al. (1995) | V | |
| | $1.4\times10^{-3}$ | | Hwang et al. (1992) | V | |
| | $1.1\times10^{-3}$ | | Yaws (2003) | X | 258 |
| | $1.1\times10^{-3}$ | | Yaws (2003) | X | 237 |
| | $1.8\times10^{-3}$ | | Dupeux et al. (2022) | Q | 259 |
| | $2.2\times10^{-2}$ | | Keshavarz et al. (2022) | Q | |
| | $4.8\times10^{-2}$ | | Duchowicz et al. (2020) | Q | 299 |
| | $6.1\times10^{-4}$ | | Gharagheizi et al. (2012) | Q | |
| | $1.7\times10^{-3}$ | | Gharagheizi et al. (2010) | Q | 246 |
| | $2.7\times10^{-3}$ | | Hilal et al. (2008) | Q | |
| | $3.4\times10^{-3}$ | | Modarresi et al. (2007) | Q | 67 |
| | | 4300 | Kühne et al. (2005) | Q | |
| | $2.2\times10^{-3}$ | | Yaffe et al. (2003) | Q | 248, 272 |
| | $3.2\times10^{-3}$ | | Yao et al. (2002) | Q | 229 |
| | $2.3\times10^{-3}$ | | English and Carroll (2001) | Q | 230, 231 |
| | $2.5\times10^{-3}$ | | Katritzky et al. (1998) | Q | |
| | $1.2\times10^{-3}$ | | Nirmalakhandan et al. (1997) | Q | |
| | $2.2\times10^{-3}$ | | Duchowicz et al. (2020) | ? | 185, 21 |
| | $1.1\times10^{-3}$ | | Bartelt-Hunt et al. (2008) | ? | 21 |
| | | 4200 | Kühne et al. (2005) | ? | |
| | $1.1\times10^{-3}$ | | Yaws et al. (2003) | ? | 21 |
| | $1.1\times10^{-3}$ | | Yaws (1999) | ? | 21 |
| | $1.1\times10^{-3}$ | | Yaws and Yang (1992) | ? | 21 |
| | $2.2\times10^{-3}$ | | Abraham et al. (1990) | ? | |



Table A9.1: Sulfur (C, H, O, N, Cl, S) (. . . continued)

| Substance Formula (Trivial Name) [CAS Registry Number] InChIKey | $H_s^{cp}$ (at $T^{\ominus}$) $\left[\dfrac{\text{mol}}{\text{m}^3\,\text{Pa}}\right]$ | $\dfrac{\text{d}\ln H_s^{cp}}{\text{d}(1/T)}$ [K] | Reference | Type | Note |
|---|---|---|---|---|---|
| 2-butanethiol | $1.3\times10^{-3}$ | | Plyasunova et al. (2004) | L | |
| $C_4H_{10}S$ | $1.4\times10^{-3}$ | | Duchowicz et al. (2020) | V | 186 |
| (*sec*-butyl mercaptan) | $1.4\times10^{-3}$ | | HSDB (2015) | V | |
| [513-53-1] | $1.9\times10^{-3}$ | | Yaws et al. (2003) | V | 802 |
| LOCHFZBWPCLPAN-UHFFFAOYSA-N | $1.9\times10^{-3}$ | | Yaws (2003) | X | 237 |
| | $1.9\times10^{-2}$ | | Duchowicz et al. (2020) | Q | |
| | $9.4\times10^{-4}$ | | Gharagheizi et al. (2012) | Q | |
| | $1.8\times10^{-3}$ | | Gharagheizi et al. (2010) | Q | 246 |
| | $1.7\times10^{-3}$ | | Modarresi et al. (2007) | Q | 67 |
| | $2.0\times10^{-3}$ | | Yao et al. (2002) | Q | 229 |
| | $1.5\times10^{-3}$ | | Yaws (1999) | ? | 21 |
| 2-methyl-1-propanethiol | $1.0\times10^{-3}$ | 3600 | Zin et al. (2016) | M | 806 |
| $C_4H_{10}S$ | $1.9\times10^{-3}$ | | Yaws et al. (2003) | V | 802 |
| (isobutyl mercaptan) | $1.9\times10^{-3}$ | | Yaws (2003) | X | 237 |
| [513-44-0] | $3.3\times10^{-4}$ | | Gharagheizi et al. (2012) | Q | |
| BDFAOUQQXJIZDG-UHFFFAOYSA-N | $2.1\times10^{-3}$ | | Gharagheizi et al. (2010) | Q | 246 |
| | $2.4\times10^{-3}$ | | Hilal et al. (2008) | Q | |
| | $2.0\times10^{-3}$ | | Yao et al. (2002) | Q | 229, 267 |
| | $1.4\times10^{-3}$ | | Yaws (1999) | ? | 21 |
| 2-methyl-2-propanethiol | $4.5\times10^{-4}$ | | Plyasunova et al. (2004) | L | |
| $C_4H_{10}S$ | $1.9\times10^{-3}$ | | Yaws et al. (2003) | V | 802 |
| (*tert*-butyl mercaptan) | $1.9\times10^{-3}$ | | Yaws (2003) | X | 237 |
| [75-66-1] | $1.6\times10^{-3}$ | | HSDB (2015) | Q | 99 |
| WMXCDAVJEZZYLT-UHFFFAOYSA-N | $1.6\times10^{-3}$ | | Gharagheizi et al. (2012) | Q | |
| | $1.9\times10^{-3}$ | | Gharagheizi et al. (2010) | Q | 246 |
| | $6.1\times10^{-4}$ | | Hilal et al. (2008) | Q | |
| | $1.1\times10^{-3}$ | | Yao et al. (2002) | Q | 229 |
| | $1.8\times10^{-3}$ | | Yaws (1999) | ? | 21 |
| 1,4-dithiane | $2.4\times10^{-1}$ | | Duchowicz et al. (2020) | V | 186 |
| $C_4H_8S_2$ | $2.3\times10^{-1}$ | | HSDB (2015) | V | |
| [505-29-3] | $6.9$ | | Duchowicz et al. (2020) | Q | |
| LOZWAPSEEHRYPG-UHFFFAOYSA-N | | | | | |
| 1-pentanethiol | $8.7\times10^{-4}$ | | Plyasunova et al. (2004) | L | |
| $C_5H_{11}SH$ | $8.2\times10^{-4}$ | | Duchowicz et al. (2020) | V | 186 |
| (pentyl mercaptan) | $8.2\times10^{-4}$ | | HSDB (2015) | V | |
| [110-66-7] | $1.4\times10^{-3}$ | | Yaws et al. (2003) | V | 802 |
| ZRKMQKLGEQPLNS-UHFFFAOYSA-N | $7.3\times10^{-4}$ | | Amoore and Buttery (1978) | V | |
| | $1.4\times10^{-3}$ | | Yaws (2003) | X | 237 |
| | $4.8\times10^{-2}$ | | Duchowicz et al. (2020) | Q | |
| | $4.6\times10^{-4}$ | | Gharagheizi et al. (2012) | Q | |
| | $1.3\times10^{-3}$ | | Gharagheizi et al. (2010) | Q | 246 |
| | $2.3\times10^{-3}$ | | Hilal et al. (2008) | Q | |
| | $2.5\times10^{-3}$ | | Modarresi et al. (2007) | Q | 67 |
| | $1.6\times10^{-3}$ | | Yao et al. (2002) | Q | 229, 267 |
| | $7.1\times10^{-4}$ | | Yaws (1999) | ? | 21 |



Table A9.1: Sulfur (C, H, O, N, Cl, S) (. . . continued)

| Substance / Formula / (Trivial Name) / [CAS Registry Number] / InChIKey | $H_s^{cp}$ (at $T^{\ominus}$) $\left[\dfrac{\text{mol}}{\text{m}^3\,\text{Pa}}\right]$ | $\dfrac{\text{d}\ln H_s^{cp}}{\text{d}(1/T)}$ [K] | Reference | Type | Note |
|---|---|---|---|---|---|
| 2,2-dimethyl-1-propanethiol | $1.5\times10^{-3}$ | | Yaws et al. (2003) | V | 802 |
| $C_5H_{12}S$ | $1.5\times10^{-3}$ | | Yaws (2003) | X | 237 |
| [1679-08-9] | $1.1\times10^{-4}$ | | Gharagheizi et al. (2012) | Q | |
| LSUXMVNABVPWMF-UHFFFAOYSA-N | $1.4\times10^{-3}$ | | Gharagheizi et al. (2010) | Q | 246 |
| 2-methyl-1-butanethiol | $1.5\times10^{-3}$ | | Yaws et al. (2003) | V | 802 |
| $C_5H_{12}S$ | $1.5\times10^{-3}$ | | Yaws (2003) | X | 237 |
| [1878-18-8] | $2.6\times10^{-4}$ | | Gharagheizi et al. (2012) | Q | |
| WGQKBCSACFQGQY-UHFFFAOYSA-N | $1.5\times10^{-3}$ | | Gharagheizi et al. (2010) | Q | 246 |
| 2-methyl-2-butanethiol | $1.5\times10^{-3}$ | | Yaws et al. (2003) | V | 802 |
| $C_5H_{12}S$ | $1.5\times10^{-3}$ | | Yaws (2003) | X | 237 |
| [1679-09-0] | $8.6\times10^{-4}$ | | Gharagheizi et al. (2012) | Q | |
| IQIBYAHJXQVQGB-UHFFFAOYSA-N | $1.5\times10^{-3}$ | | Gharagheizi et al. (2010) | Q | 246 |
| | $8.9\times10^{-4}$ | | Yao et al. (2002) | Q | 229, 807 |
| | $1.0\times10^{-3}$ | | Yaws (1999) | ? | 21 |
| 2-pentanethiol | $1.5\times10^{-3}$ | | Yaws et al. (2003) | V | 802 |
| $C_5H_{12}S$ | $1.5\times10^{-3}$ | | Yaws (2003) | X | 258 |
| [2084-19-7] | $1.5\times10^{-3}$ | | Yaws (2003) | X | 237 |
| QUSTYFNPKBDELJ-UHFFFAOYSA-N | $1.0\times10^{-3}$ | | Dupeux et al. (2022) | Q | 259 |
| | $6.3\times10^{-4}$ | | Gharagheizi et al. (2012) | Q | |
| | $1.3\times10^{-3}$ | | Gharagheizi et al. (2010) | Q | 246 |
| 3-pentanethiol | $1.5\times10^{-3}$ | | Yaws et al. (2003) | V | 802 |
| $C_5H_{12}S$ | $1.5\times10^{-3}$ | | Yaws (2003) | X | 237 |
| [616-31-9] | $4.5\times10^{-4}$ | | Gharagheizi et al. (2012) | Q | |
| WICKAMSPKJXSGN-UHFFFAOYSA-N | $1.4\times10^{-3}$ | | Gharagheizi et al. (2010) | Q | 246 |
| 3-methyl-1-butanethiol | $9.5\times10^{-4}$ | | Plyasunova et al. (2004) | L | |
| $C_5H_{12}S$ | $1.5\times10^{-3}$ | | Yaws et al. (2003) | V | 802 |
| [541-31-1] | $1.5\times10^{-3}$ | | Yaws (2003) | X | 237 |
| GIJGXNFNUUFEGH-UHFFFAOYSA-N | $3.7\times10^{-4}$ | | Gharagheizi et al. (2012) | Q | |
| | $1.4\times10^{-3}$ | | Gharagheizi et al. (2010) | Q | 246 |
| 3-methyl-2-butanethiol | $1.5\times10^{-3}$ | | Yaws et al. (2003) | V | 802 |
| $C_5H_{12}S$ | $1.5\times10^{-3}$ | | Yaws (2003) | X | 237 |
| [2084-18-6] | $4.0\times10^{-4}$ | | Gharagheizi et al. (2012) | Q | |
| BFLXFRNPNMTTAA-UHFFFAOYSA-N | $1.5\times10^{-3}$ | | Gharagheizi et al. (2010) | Q | 246 |
| 1-hexanethiol | $5.2\times10^{-4}$ | | Plyasunova et al. (2004) | L | |
| $C_6H_{14}S$ | $1.1\times10^{-3}$ | | Yaws et al. (2003) | V | 802 |
| (hexyl mercaptan) | $1.1\times10^{-3}$ | | Yaws (2003) | X | 237 |
| [111-31-9] | $3.5\times10^{-4}$ | | Gharagheizi et al. (2012) | Q | |
| PMBXCGGQNSVESQ-UHFFFAOYSA-N | $1.0\times10^{-3}$ | | Gharagheizi et al. (2010) | Q | 246 |
| | $1.9\times10^{-3}$ | | Hilal et al. (2008) | Q | |
| | $1.4\times10^{-3}$ | | Yao et al. (2002) | Q | 229 |
| | $4.5\times10^{-4}$ | | Yaws (1999) | ? | 21 |



Table A9.1: Sulfur (C, H, O, N, Cl, S) (... continued)

| Substance Formula (Trivial Name) [CAS Registry Number] InChIKey | $H_s^{cp}$ (at $T^{\ominus}$) $\left[\dfrac{\text{mol}}{\text{m}^3\,\text{Pa}}\right]$ | $\dfrac{\mathrm{d}\ln H_s^{cp}}{\mathrm{d}(1/T)}$ [K] | Reference | Type | Note |
|---|---|---|---|---|---|
| 2-hexanethiol | $1.2\times10^{-3}$ | | Yaws et al. (2003) | V | 802 |
| $C_6H_{14}S$ | $1.2\times10^{-3}$ | | Yaws (2003) | X | 237 |
| [1679-06-7] | $4.9\times10^{-4}$ | | Gharagheizi et al. (2012) | Q | |
| ABNPJVOPTXYSQW-UHFFFAOYSA-N | $1.0\times10^{-3}$ | | Gharagheizi et al. (2010) | Q | 246 |
| 1-heptanethiol | $4.2\times10^{-4}$ | | Plyasunova et al. (2004) | L | |
| $C_7H_{16}S$ | $4.0\times10^{-4}$ | | Duchowicz et al. (2020) | V | 186 |
| (heptyl mercaptan) | $9.5\times10^{-4}$ | | Yaws et al. (2003) | V | 802 |
| [1639-09-4] | $8.2\times10^{-4}$ | | Yaws (2003) | X | 237 |
| VPIAKHNXCOTPAY-UHFFFAOYSA-N | $4.8\times10^{-2}$ | | Duchowicz et al. (2020) | Q | |
| | $2.7\times10^{-4}$ | | Gharagheizi et al. (2012) | Q | |
| | $8.4\times10^{-4}$ | | Gharagheizi et al. (2010) | Q | 246 |
| | $2.7\times10^{-3}$ | | Hilal et al. (2008) | Q | |
| | $1.2\times10^{-3}$ | | Yao et al. (2002) | Q | 229, 267 |
| | $3.3\times10^{-4}$ | | Yaws (1999) | ? | 21 |
| 2-heptanethiol | $1.3\times10^{-3}$ | | Yaws et al. (2003) | V | 802 |
| $C_7H_{16}S$ | $8.2\times10^{-4}$ | | Yaws (2003) | X | 237 |
| [628-00-2] | $4.1\times10^{-4}$ | | Gharagheizi et al. (2012) | Q | |
| DAZNOIJJVKASGS-UHFFFAOYSA-N | $8.3\times10^{-4}$ | | Gharagheizi et al. (2010) | Q | 246 |
| 1-octanethiol | $7.1\times10^{-4}$ | | Yaws et al. (2003) | V | 802 |
| $C_8H_{18}S$ | $6.9\times10^{-4}$ | | Yaws (2003) | X | 237 |
| (octyl mercaptan) | $4.3\times10^{-4}$ | | HSDB (2015) | Q | 99 |
| [111-88-6] | $2.0\times10^{-4}$ | | Gharagheizi et al. (2012) | Q | |
| KZCOBXFFBQJQHH-UHFFFAOYSA-N | $7.4\times10^{-4}$ | | Gharagheizi et al. (2010) | Q | 246 |
| | $1.3\times10^{-3}$ | | Hilal et al. (2008) | Q | |
| | $1.1\times10^{-3}$ | | Yao et al. (2002) | Q | 229 |
| | $3.0\times10^{-4}$ | | Yaws (1999) | ? | 21 |
| *tert*-octanethiol | $2.0\times10^{-3}$ | | Yaws et al. (2003) | V | 802 |
| $C_8H_{18}S$ | $6.2\times10^{-4}$ | | Yaws (2003) | X | 237 |
| (*tert*-octyl mercaptan) | $5.2\times10^{-4}$ | | HSDB (2015) | Q | 99 |
| [141-59-3] | $2.8\times10^{-4}$ | | Gharagheizi et al. (2012) | Q | |
| QZLAEIZEPJAELS-UHFFFAOYSA-N | $6.6\times10^{-4}$ | | Gharagheizi et al. (2010) | Q | 246 |
| | $4.2\times10^{-4}$ | | Yao et al. (2002) | Q | 229 |
| | $2.6\times10^{-4}$ | | Yaws (1999) | ? | 21 |
| 2-octanethiol | $9.9\times10^{-4}$ | | Yaws et al. (2003) | V | 802 |
| $C_8H_{18}S$ | $6.6\times10^{-4}$ | | Yaws (2003) | X | 237 |
| [3001-66-9] | $3.0\times10^{-4}$ | | Gharagheizi et al. (2012) | Q | |
| BZXFEMZFRLXGCY-UHFFFAOYSA-N | $6.9\times10^{-4}$ | | Gharagheizi et al. (2010) | Q | 246 |
| 1-nonanethiol | $5.7\times10^{-4}$ | | Yaws et al. (2003) | V | 802 |
| $C_9H_{20}S$ | $6.7\times10^{-4}$ | | Yaws (2003) | X | 237 |
| (nonyl mercaptan) | $1.5\times10^{-4}$ | | Gharagheizi et al. (2012) | Q | |
| [1455-21-6] | $7.1\times10^{-4}$ | | Gharagheizi et al. (2010) | Q | 246 |
| ZVEZMVFBMOOHAT-UHFFFAOYSA-N | $1.2\times10^{-3}$ | | Hilal et al. (2008) | Q | |
| | $8.9\times10^{-4}$ | | Yao et al. (2002) | Q | 229, 267 |
| | $3.7\times10^{-4}$ | | Yaws (1999) | ? | 21 |



Table A9.1: Sulfur (C, H, O, N, Cl, S) (. . . continued)

| Substance Formula (Trivial Name) [CAS Registry Number] InChIKey | $H_s^{cp}$ (at $T^{\ominus}$) $\left[\dfrac{\text{mol}}{\text{m}^3\,\text{Pa}}\right]$ | $\dfrac{\text{d}\ln H_s^{cp}}{\text{d}(1/T)}$ [K] | Reference | Type | Note |
|---|---|---|---|---|---|
| 2-nonanethiol | $7.7{\times}10^{-4}$ | | Yaws et al. (2003) | V | 802 |
| $C_9H_{20}S$ | $5.1{\times}10^{-4}$ | | Yaws (2003) | X | 237 |
| [13281-11-3] | $2.3{\times}10^{-4}$ | | Gharagheizi et al. (2012) | Q | |
| UOMSUBPWUZCGQU-UHFFFAOYSA-N | $6.1{\times}10^{-4}$ | | Gharagheizi et al. (2010) | Q | 246 |
| 1-decanethiol | $5.0{\times}10^{-4}$ | | Yaws et al. (2003) | V | 802 |
| $C_{10}H_{22}S$ | $8.1{\times}10^{-4}$ | | Yaws (2003) | X | 237 |
| (decyl mercaptan) | $1.6{\times}10^{-4}$ | | Gharagheizi et al. (2012) | Q | |
| [143-10-2] | $7.7{\times}10^{-4}$ | | Gharagheizi et al. (2010) | Q | 246 |
| VTXVGVNLYGSIAR-UHFFFAOYSA-N | $9.9{\times}10^{-4}$ | | Hilal et al. (2008) | Q | |
| | $8.2{\times}10^{-4}$ | | Yao et al. (2002) | Q | 229 |
| | $6.5{\times}10^{-4}$ | | Yaws (1999) | ? | 21 |
| 2-decanethiol | $6.5{\times}10^{-4}$ | | Yaws et al. (2003) | V | 802 |
| $C_{10}H_{22}S$ | $4.4{\times}10^{-4}$ | | Yaws (2003) | X | 237 |
| [13402-60-3] | $2.5{\times}10^{-4}$ | | Gharagheizi et al. (2012) | Q | |
| NWKXKAHGQAWFQP-UHFFFAOYSA-N | $5.8{\times}10^{-4}$ | | Gharagheizi et al. (2010) | Q | 246 |
| 1-undecanethiol | $4.9{\times}10^{-4}$ | | Yaws et al. (2003) | V | 802 |
| $C_{11}H_{24}S$ | $2.6{\times}10^{-3}$ | | Yaws (2003) | X | 237 |
| (undecyl mercaptan) | $9.3{\times}10^{-4}$ | | Gharagheizi et al. (2010) | Q | 246 |
| [5332-52-5] | $6.7{\times}10^{-4}$ | | Yao et al. (2002) | Q | 229 |
| CCIDWXHLGNEQSL-UHFFFAOYSA-N | $3.4{\times}10^{-3}$ | | Yaws (1999) | ? | 21 |
| 2-undecanethiol | $6.1{\times}10^{-4}$ | | Yaws et al. (2003) | V | 802 |
| $C_{11}H_{24}S$ | $6.6{\times}10^{-4}$ | | Yaws (2003) | X | 237 |
| [62155-02-6] | $2.9{\times}10^{-4}$ | | Gharagheizi et al. (2012) | Q | |
| KRMLVHZORKTOLI-UHFFFAOYSA-N | $6.0{\times}10^{-4}$ | | Gharagheizi et al. (2010) | Q | 246 |
| 1-dodecanethiol | $5.3{\times}10^{-4}$ | | Yaws et al. (2003) | V | 802 |
| $C_{12}H_{26}S$ | $5.8{\times}10^{-4}$ | | Yaws (2003) | X | 237 |
| (dodecyl mercaptan) | $1.7{\times}10^{-4}$ | | HSDB (2015) | Q | 99 |
| [112-55-0] | $1.9{\times}10^{-4}$ | | Gharagheizi et al. (2012) | Q | |
| WNAHIZMDSQCWRP-UHFFFAOYSA-N | $1.3{\times}10^{-3}$ | | Gharagheizi et al. (2010) | Q | 246 |
| | $5.3{\times}10^{-4}$ | | Yao et al. (2002) | Q | 229 |
| | $1.4{\times}10^{-3}$ | | Yaws (1999) | ? | 21 |
| 2-dodecanethiol | $6.3{\times}10^{-4}$ | | Yaws et al. (2003) | V | 802 |
| $C_{12}H_{26}S$ | $7.2{\times}10^{-4}$ | | Yaws (2003) | X | 237 |
| [14402-50-7] | $3.0{\times}10^{-4}$ | | Gharagheizi et al. (2012) | Q | |
| UROXMPKAGAWKPP-UHFFFAOYSA-N | $6.9{\times}10^{-4}$ | | Gharagheizi et al. (2010) | Q | 246 |
| 5-propyl-5-nonanethiol | $1.7{\times}10^{-4}$ | | Zhang et al. (2010) | Q | 287, 288 |
| $C_{12}H_{26}S$ | $5.8{\times}10^{-4}$ | | Zhang et al. (2010) | Q | 287, 289 |
| BGQHJKADBZPYGJ-UHFFFAOYSA-N | $2.4{\times}10^{-3}$ | | Zhang et al. (2010) | Q | 287, 290 |
| | $9.7{\times}10^{-5}$ | | Zhang et al. (2010) | Q | 287, 291 |





Table A9.1: Sulfur (C, H, O, N, Cl, S) (...continued)

| Substance<br>Formula<br>(Trivial Name)<br>[CAS Registry Number]<br>InChIKey | $H_s^{cp}$<br>(at $T^{\ominus}$)<br>$\left[\dfrac{\mathrm{mol}}{\mathrm{m}^3\,\mathrm{Pa}}\right]$ | $\dfrac{\mathrm{d}\ln H_s^{cp}}{\mathrm{d}(1/T)}$<br>[K] | Reference | Type | Note |
|---|---|---|---|---|---|
| 1-tridecanethiol<br>$C_{13}H_{28}S$<br>[19484-26-5]<br>IPBROXKVGHZHJV-UHFFFAOYSA-N | $6.4\times10^{-4}$<br>$2.8\times10^{-3}$<br>$2.3\times10^{-3}$ | | Yaws et al. (2003)<br>Yaws (2003)<br>Gharagheizi et al. (2010) | V<br>X<br>Q | 802<br>237<br>246 |
| 2-tridecanethiol<br>$C_{13}H_{28}S$<br>[62155-03-7]<br>ZAPRCDVCNIREGW-UHFFFAOYSA-N | $7.3\times10^{-4}$<br>$9.6\times10^{-4}$<br>$3.2\times10^{-4}$<br>$9.0\times10^{-4}$ | | Yaws et al. (2003)<br>Yaws (2003)<br>Gharagheizi et al. (2012)<br>Gharagheizi et al. (2010) | V<br>X<br>Q<br>Q | 802<br>237<br><br>246 |
| 1-tetradecanethiol<br>$C_{14}H_{30}S$<br>[2079-95-0]<br>GEKDEMKPCKTKEC-UHFFFAOYSA-N | $8.5\times10^{-4}$<br>$5.1\times10^{-3}$<br>$4.9\times10^{-3}$ | | Yaws et al. (2003)<br>Yaws (2003)<br>Gharagheizi et al. (2010) | V<br>X<br>Q | 802<br>237<br>246 |
| 2-tetradecanethiol<br>$C_{14}H_{30}S$<br>[62155-04-8]<br>YESAYZVWGLMPDY-UHFFFAOYSA-N | $9.2\times10^{-4}$<br>$1.5\times10^{-3}$<br>$3.4\times10^{-4}$<br>$1.4\times10^{-3}$ | | Yaws et al. (2003)<br>Yaws (2003)<br>Gharagheizi et al. (2012)<br>Gharagheizi et al. (2010) | V<br>X<br>Q<br>Q | 802<br>237<br><br>246 |
| dimethyl sulfide<br>$CH_3SCH_3$<br>(DMS)<br>[75-18-3]<br>QMMFVYPAHWMCMS-UHFFFAOYSA-N | $5.3\times10^{-3}$<br>$5.3\times10^{-3}$<br>$5.4\times10^{-3}$<br>$5.6\times10^{-3}$<br>$5.3\times10^{-3}$<br>$5.3\times10^{-3}$<br>$5.3\times10^{-3}$<br>$5.2\times10^{-3}$<br>$5.3\times10^{-3}$<br>$4.7\times10^{-3}$<br>$4.6\times10^{-3}$<br>$4.8\times10^{-3}$<br>$5.2\times10^{-3}$<br>$5.5\times10^{-3}$<br>$4.9\times10^{-3}$<br>$6.4\times10^{-3}$<br>$4.9\times10^{-3}$<br>$6.4\times10^{-3}$<br>$7.2\times10^{-3}$<br>$4.7\times10^{-3}$<br>$4.9\times10^{-3}$<br>$1.6\times10^{-2}$<br>$4.2\times10^{-3}$<br>$4.7\times10^{-3}$<br>$5.7\times10^{-3}$<br>$5.5\times10^{-3}$<br>$5.6\times10^{-3}$<br>$6.1\times10^{-3}$<br>$1.6\times10^{-3}$ | 3500<br>3500<br>3500<br>3500<br>3500<br>3500<br>3800<br>3600<br>3500<br>4700<br><br>2800<br>3600<br>3800<br><br>4100<br><br><br><br>3700<br><br><br>4300<br>3100<br>2700<br>3500<br>4000<br><br> | Burkholder et al. (2019)<br>Burkholder et al. (2015)<br>Brockbank (2013)<br>Warneck and Williams (2012)<br>Sander et al. (2011)<br>Sander et al. (2006)<br>Plyasunova et al. (2004)<br>Fogg and Sangster (2003)<br>Staudinger and Roberts (2001)<br>Bruneel et al. (2016)<br>Schuhfried et al. (2011)<br>Falabella (2007)<br>Coquelet and Richon (2005)<br>Iliuta and Larachi (2005a)<br>Straver and de Loos (2005)<br>Barcellos da Rosa et al. (2003)<br>Pollien et al. (2003)<br>van Ruth et al. (2002)<br>van Ruth and Villeneuve (2002)<br>Gershenzon et al. (2001)<br>van Ruth et al. (2001)<br>Marin et al. (1999)<br>Wong and Wang (1997)<br>De Bruyn et al. (1995b)<br>Tsuji et al. (1990)<br>Dacey et al. (1984)<br>Przyjazny et al. (1983)<br>Vitenberg et al. (1975)<br>Lovelock et al. (1972) | L<br>L<br>L<br>L<br>L<br>L<br>L<br>L<br>L<br>M<br>M<br>M<br>M<br>M<br>M<br>M<br>M<br>M<br>M<br>M<br>M<br>M<br>M<br>M<br>M<br>M<br>M<br>M<br>M | <br><br>1<br><br><br><br><br><br><br><br><br>11, 338<br><br><br><br><br><br>14<br>14, 361<br><br>14<br><br><br><br>62<br><br><br>12<br> |





Table A9.1: Sulfur (C, H, O, N, Cl, S) (...continued)

| Substance<br>Formula<br>(Trivial Name)<br>[CAS Registry Number]<br>InChIKey | $H_s^{cp}$<br>(at $T^{\ominus}$)<br>$\left[\dfrac{\text{mol}}{\text{m}^3\,\text{Pa}}\right]$ | $\dfrac{\text{d}\ln H_s^{cp}}{\text{d}(1/T)}$<br><br>[K] | Reference | Type | Note |
|---|---|---|---|---|---|
| | | | Mackay et al. (2006d) | V | 558 |
| | $4.2\times10^{-3}$ | | Marin et al. (1999) | V | |
| | $1.3\times10^{-1}$ | | Mackay et al. (1995) | V | |
| | $5.4\times10^{-3}$ | | Hine and Mookerjee (1975) | V | |
| | $5.5\times10^{-3}$ | | Hine and Weimar (1965) | V | |
| | $7.0\times10^{-3}$ | | Vitenberg et al. (1975) | R | 12 |
| | $6.0\times10^{-3}$ | 3700 | Bagno et al. (1991) | T | 473 |
| | $5.5\times10^{-3}$ | | Yaws (2003) | X | 237 |
| | $6.1\times10^{-3}$ | | Gaffney and Senum (1984) | X | 389 |
| | $4.4\times10^{-3}$ | | Cline and Bates (1983) | C | 70 |
| | $2.3\times10^{-3}$ | | Keshavarz et al. (2022) | Q | |
| | $1.7\times10^{-1}$ | | Duchowicz et al. (2020) | Q | 184 |
| | $1.7\times10^{-3}$ | | Wang et al. (2017) | Q | 80, 238 |
| | $1.7\times10^{-2}$ | | Wang et al. (2017) | Q | 80, 239 |
| | $4.9\times10^{-3}$ | | Wang et al. (2017) | Q | 80, 240 |
| | $1.9\times10^{-3}$ | | Gharagheizi et al. (2012) | Q | |
| | $8.7\times10^{-3}$ | | Gharagheizi et al. (2010) | Q | 246 |
| | $1.2\times10^{-2}$ | | Hilal et al. (2008) | Q | |
| | $3.3\times10^{-3}$ | | Modarresi et al. (2007) | Q | 67 |
| | $7.2\times10^{-3}$ | | Hertel et al. (2007) | Q | 467 |
| | | 3100 | Kühne et al. (2005) | Q | |
| | $6.2\times10^{-3}$ | | Yaffe et al. (2003) | Q | 248, 249 |
| | $4.8\times10^{-3}$ | | English and Carroll (2001) | Q | 230, 231 |
| | $5.0\times10^{-3}$ | | Marin et al. (1999) | Q | |
| | $3.1\times10^{-3}$ | | Katritzky et al. (1998) | Q | |
| | $6.5\times10^{-3}$ | | Nirmalakhandan et al. (1997) | Q | |
| | $7.9\times10^{-4}$ | | Russell et al. (1992) | Q | 279 |
| | $6.4\times10^{-3}$ | | Suzuki et al. (1992) | Q | 232 |
| | $6.1\times10^{-3}$ | | Duchowicz et al. (2020) | ? | 185, 21 |
| | | 3500 | Kühne et al. (2005) | ? | |
| | $5.5\times10^{-3}$ | | Yaws et al. (2003) | ? | 21 |
| | $4.9\times10^{-3}$ | | Yaws (1999) | ? | 21 |
| | $1.7\times10^{-3}$ | | Abraham et al. (1990) | ? | |
| ethyl methyl sulfide<br>$C_2H_5SCH_3$<br>[624-89-5]<br>WXEHBUMAEPOYKP-UHFFFAOYSA-N | $4.2\times10^{-3}$ | | Burkholder et al. (2019) | L | |
| | $4.2\times10^{-3}$ | | Burkholder et al. (2015) | L | |
| | $4.2\times10^{-3}$ | | Schuhfried et al. (2011) | M | |
| | $5.4\times10^{-3}$ | | Yaws et al. (2003) | V | 802 |
| | $5.1\times10^{-3}$ | | Bagno et al. (1991) | T | 473 |
| | $5.4\times10^{-3}$ | | Yaws (2003) | X | 237 |
| | $2.0\times10^{-3}$ | | Gharagheizi et al. (2012) | Q | |
| | $5.3\times10^{-3}$ | | Gharagheizi et al. (2010) | Q | 246 |
| | $8.6\times10^{-3}$ | | Hilal et al. (2008) | Q | |
| | $5.1\times10^{-3}$ | | Yao et al. (2002) | Q | 229 |
| | $3.5\times10^{-3}$ | | English and Carroll (2001) | Q | 230, 231 |
| | $4.4\times10^{-3}$ | | Nirmalakhandan et al. (1997) | Q | |
| | $5.0\times10^{-3}$ | | Yaws (1999) | ? | 21 |



Table A9.1: Sulfur (C, H, O, N, Cl, S) (…continued)

| Substance<br>Formula<br>(Trivial Name)<br>[CAS Registry Number]<br>InChIKey | $H_s^{cp}$<br>(at $T^{\ominus}$)<br>$\left[\dfrac{\text{mol}}{\text{m}^3\,\text{Pa}}\right]$ | $\dfrac{\text{d}\ln H_s^{cp}}{\text{d}(1/T)}$<br><br>[K] | Reference | Type | Note |
|---|---|---|---|---|---|
| diethyl sulfide<br>$C_2H_5SC_2H_5$<br>[352-93-2]<br>LJSQFQKUNVCTIA-UHFFFAOYSA-N | $3.5\times10^{-3}$ | | Burkholder et al. (2019) | L | |
| | $3.5\times10^{-3}$ | | Burkholder et al. (2015) | L | |
| | $4.7\times10^{-3}$ | 4800 | Plyasunova et al. (2004) | L | |
| | $3.5\times10^{-3}$ | | Schuhfried et al. (2011) | M | |
| | $5.4\times10^{-3}$ | 4900 | Przyjazny et al. (1983) | M | |
| | $5.1\times10^{-1}$ | | Mackay et al. (2006d) | V | |
| | $4.5\times10^{-3}$ | | Hine and Mookerjee (1975) | V | |
| | $4.5\times10^{-3}$ | | Yaws (2003) | X | 237 |
| | $4.3\times10^{-3}$ | | Keshavarz et al. (2022) | Q | |
| | $2.0\times10^{-2}$ | | Duchowicz et al. (2020) | Q | 184 |
| | $3.4\times10^{-3}$ | | Gharagheizi et al. (2012) | Q | |
| | $3.7\times10^{-3}$ | | Gharagheizi et al. (2010) | Q | 246 |
| | $6.0\times10^{-3}$ | | Hilal et al. (2008) | Q | |
| | $3.6\times10^{-3}$ | | Modarresi et al. (2007) | Q | 67 |
| | $4.8\times10^{-3}$ | | Yaffe et al. (2003) | Q | 248, 249 |
| | $4.0\times10^{-3}$ | | Yao et al. (2002) | Q | 229 |
| | $4.6\times10^{-3}$ | | English and Carroll (2001) | Q | 230, 274 |
| | $2.6\times10^{-3}$ | | Katritzky et al. (1998) | Q | |
| | $2.9\times10^{-3}$ | | Nirmalakhandan et al. (1997) | Q | |
| | $1.5\times10^{-3}$ | | Russell et al. (1992) | Q | 279 |
| | $3.9\times10^{-3}$ | | Suzuki et al. (1992) | Q | 232 |
| | $5.9\times10^{-3}$ | | Duchowicz et al. (2020) | ? | 185, 21 |
| | $1.1\times10^{-2}$ | | Yaws et al. (2003) | ? | 21 |
| | $4.4\times10^{-3}$ | | Yaws (1999) | ? | 21, 12 |
| | $5.7\times10^{-3}$ | | Yaws and Yang (1992) | ? | 21, 12 |
| | $4.7\times10^{-3}$ | | Abraham et al. (1990) | ? | |
| methyl isopropyl sulfide<br>$C_4H_{10}S$<br>[1551-21-9]<br>ROSSIHMZZJOVOU-UHFFFAOYSA-N | $4.5\times10^{-3}$ | | Yaws et al. (2003) | V | 802 |
| | $4.5\times10^{-3}$ | | Yaws (2003) | X | 237 |
| | $2.7\times10^{-3}$ | | Gharagheizi et al. (2012) | Q | |
| | $4.6\times10^{-3}$ | | Gharagheizi et al. (2010) | Q | 246 |
| | $2.6\times10^{-3}$ | | Yao et al. (2002) | Q | 229 |
| | $3.5\times10^{-3}$ | | Yaws (1999) | ? | 21 |
| methyl propyl sulfide<br>$C_4H_{10}S$<br>[3877-15-4]<br>ZOASGOXWEHUTKZ-UHFFFAOYSA-N | $4.3\times10^{-3}$ | | Plyasunova et al. (2004) | L | |
| | $4.3\times10^{-3}$ | | Mazza (1980) | M | |
| | $4.5\times10^{-3}$ | | Yaws et al. (2003) | V | 802 |
| | $4.5\times10^{-3}$ | | Yaws (2003) | X | 237 |
| | $1.1\times10^{-3}$ | | Gharagheizi et al. (2012) | Q | |
| | $4.2\times10^{-3}$ | | Gharagheizi et al. (2010) | Q | 246 |
| | $4.1\times10^{-3}$ | | Yao et al. (2002) | Q | 229 |
| | $3.1\times10^{-3}$ | | Yaws (1999) | ? | 21 |
| tetrahydrothiophene<br>$C_4H_8S$<br>[110-01-0]<br>RAOIDOHSFRTOEL-UHFFFAOYSA-N | $2.2\times10^{-3}$ | | Yaws et al. (2003) | V | 802 |
| | $1.7\times10^{-2}$ | | Gharagheizi et al. (2012) | Q | |



Table A9.1: Sulfur (C, H, O, N, Cl, S) (...continued)

| Substance Formula (Trivial Name) [CAS Registry Number] InChIKey | $H_s^{cp}$ (at $T^{\ominus}$) $\left[\dfrac{\mathrm{mol}}{\mathrm{m^3\,Pa}}\right]$ | $\dfrac{\mathrm{d\ln} H_s^{cp}}{\mathrm{d}(1/T)}$ [K] | Reference | Type | Note |
|---|---|---|---|---|---|
| 2-methyl-3-thiapentane C$_5$H$_{12}$S [5145-99-3] NZUQQADVSXWVNW-UHFFFAOYSA-N | $3.7\times10^{-3}$ $3.7\times10^{-3}$ $3.7\times10^{-3}$ $3.3\times10^{-3}$ | | Yaws et al. (2003) Yaws (2003) Gharagheizi et al. (2012) Gharagheizi et al. (2010) | V X Q Q | 802 237 246 |
| 3,3-dimethyl-2-thiabutane C$_5$H$_{12}$S [6163-64-0] CJFVCTVYZFTORU-UHFFFAOYSA-N | $3.6\times10^{-3}$ $3.6\times10^{-3}$ $2.9\times10^{-3}$ $3.9\times10^{-3}$ | | Yaws et al. (2003) Yaws (2003) Gharagheizi et al. (2012) Gharagheizi et al. (2010) | V X Q Q | 802 237 246 |
| 3-methyl-2-thiapentane C$_5$H$_{12}$S [10359-64-5] IJRCRFQMYAJPPO-UHFFFAOYSA-N | $3.6\times10^{-3}$ $3.5\times10^{-3}$ $1.3\times10^{-3}$ $3.6\times10^{-3}$ | | Yaws et al. (2003) Yaws (2003) Gharagheizi et al. (2012) Gharagheizi et al. (2010) | V X Q Q | 802 237 246 |
| 4-methyl-2-thiapentane C$_5$H$_{12}$S [5008-69-5] UYVGFIKOUAFDOZ-UHFFFAOYSA-N | $3.5\times10^{-3}$ $3.5\times10^{-3}$ $4.3\times10^{-4}$ $4.2\times10^{-3}$ | | Yaws et al. (2003) Yaws (2003) Gharagheizi et al. (2012) Gharagheizi et al. (2010) | V X Q Q | 802 237 246 |
| ethyl propyl sulfide C$_5$H$_{12}$S [4110-50-3] ZDDDFDQTSXYYSE-UHFFFAOYSA-N | $3.5\times10^{-3}$ $3.5\times10^{-3}$ $1.6\times10^{-3}$ $2.9\times10^{-3}$ $2.1\times10^{-3}$ $1.9\times10^{-3}$ | | Yaws et al. (2003) Yaws (2003) Gharagheizi et al. (2012) Gharagheizi et al. (2010) Yao et al. (2002) Yaws (1999) | V X Q Q Q ? | 802 237 246 229 21 |
| methyl butyl sulfide C$_5$H$_{12}$S [628-29-5] WCXXISMIJBRDQK-UHFFFAOYSA-N | $3.4\times10^{-3}$ $3.4\times10^{-3}$ $8.6\times10^{-4}$ $3.0\times10^{-3}$ $2.2\times10^{-3}$ $1.8\times10^{-3}$ | | Yaws et al. (2003) Yaws (2003) Gharagheizi et al. (2012) Gharagheizi et al. (2010) Yao et al. (2002) Yaws (1999) | V X Q Q Q ? | 802 237 246 229 21 |
| dipropyl sulfide C$_3$H$_7$SC$_3$H$_7$ [111-47-7] ZERULLAPCVRMCO-UHFFFAOYSA-N | $3.3\times10^{-3}$ $3.3\times10^{-3}$ $2.7\times10^{-3}$ $2.7\times10^{-3}$ $7.8\times10^{-3}$ $2.0\times10^{-2}$ $8.9\times10^{-4}$ $2.3\times10^{-3}$ $3.4\times10^{-3}$ $4.8\times10^{-3}$ $1.9\times10^{-3}$ $3.1\times10^{-3}$ $1.8\times10^{-3}$ $3.0\times10^{-3}$ | 5700 4500 4500 4500 4500 | Plyasunova et al. (2004) Przyjazny et al. (1983) Yaws et al. (2003) Yaws (2003) Keshavarz et al. (2022) Duchowicz et al. (2020) Gharagheizi et al. (2012) Gharagheizi et al. (2010) Hilal et al. (2008) Kühne et al. (2005) Yaffe et al. (2003) Yao et al. (2002) English and Carroll (2001) Nirmalakhandan et al. (1997) Duchowicz et al. (2020) Kühne et al. (2005) | L M V X Q Q Q Q Q Q Q Q Q Q ? ? | 802 237 299 246 248, 272 229 230, 231 185, 21 |



Table A9.1: Sulfur (C, H, O, N, Cl, S) (...continued)

| Substance Formula (Trivial Name) [CAS Registry Number] InChIKey | $H_s^{cp}$ (at $T^\ominus$) $\left[\dfrac{\mathrm{mol}}{\mathrm{m^3\,Pa}}\right]$ | $\dfrac{\mathrm{d}\ln H_s^{cp}}{\mathrm{d}(1/T)}$ [K] | Reference | Type | Note |
|---|---|---|---|---|---|
| | $1.2\times10^{-3}$ | | Yaws (1999) | ? | 21 |
| | $3.5\times10^{-3}$ | | Abraham et al. (1990) | ? | |
| di-(2-propyl)-sulfide | $2.8\times10^{-3}$ | 4900 | Brockbank (2013) | L | 1 |
| $(C_3H_7)_2S$ | $3.0\times10^{-3}$ | | Plyasunova et al. (2004) | L | |
| (diisopropyl sulfide) | $3.0\times10^{-3}$ | 5000 | Przyjazny et al. (1983) | M | |
| [625-80-9] | $2.9\times10^{-3}$ | | Yaws et al. (2003) | V | 802 |
| XYWDPYKBIRQXQS-UHFFFAOYSA-N | $2.9\times10^{-3}$ | | Yaws (2003) | X | 237 |
| | $7.8\times10^{-3}$ | | Keshavarz et al. (2022) | Q | |
| | $3.1\times10^{-3}$ | | Duchowicz et al. (2020) | Q | |
| | $4.8\times10^{-3}$ | | Gharagheizi et al. (2012) | Q | |
| | $3.2\times10^{-3}$ | | Gharagheizi et al. (2010) | Q | 246 |
| | $1.6\times10^{-3}$ | | Hilal et al. (2008) | Q | |
| | | 4500 | Kühne et al. (2005) | Q | |
| | $3.1\times10^{-3}$ | | Yaffe et al. (2003) | Q | 248, 249 |
| | $1.1\times10^{-3}$ | | Yao et al. (2002) | Q | 229 |
| | $1.2\times10^{-3}$ | | Nirmalakhandan et al. (1997) | Q | |
| | $3.0\times10^{-3}$ | | Duchowicz et al. (2020) | ? | 185, 21 |
| | | 4200 | Kühne et al. (2005) | ? | |
| | $1.5\times10^{-3}$ | | Yaws (1999) | ? | 21 |
| | $3.1\times10^{-3}$ | | Abraham et al. (1990) | ? | |
| 2,2-dimethyl-3-thiapentane | $2.8\times10^{-3}$ | | Yaws et al. (2003) | V | 802 |
| $C_6H_{14}S$ | $2.8\times10^{-3}$ | | Yaws (2003) | X | 237 |
| [14290-92-7] | $4.9\times10^{-3}$ | | Gharagheizi et al. (2012) | Q | |
| GZJUDUMQICJSFJ-UHFFFAOYSA-N | $2.8\times10^{-3}$ | | Gharagheizi et al. (2010) | Q | 246 |
| 2-methyl-3-thiahexane | $2.8\times10^{-3}$ | | Yaws et al. (2003) | V | 802 |
| $C_6H_{14}S$ | $2.7\times10^{-3}$ | | Yaws (2003) | X | 237 |
| [5008-73-1] | $2.1\times10^{-3}$ | | Gharagheizi et al. (2012) | Q | |
| BDFDQOJJDDORSR-UHFFFAOYSA-N | $2.6\times10^{-3}$ | | Gharagheizi et al. (2010) | Q | 246 |
| 3,3-dimethyl-2-thiapentane | $2.7\times10^{-3}$ | | Yaws et al. (2003) | V | 802 |
| $C_6H_{14}S$ | $2.7\times10^{-3}$ | | Yaws (2003) | X | 237 |
| [13286-92-5] | $1.9\times10^{-3}$ | | Gharagheizi et al. (2012) | Q | |
| SJOHDSPLOVBXOF-UHFFFAOYSA-N | $3.0\times10^{-3}$ | | Gharagheizi et al. (2010) | Q | 246 |
| 3,4-dimethyl-2-thiapentane | $2.9\times10^{-3}$ | | Yaws et al. (2003) | V | 802 |
| $C_6H_{14}S$ | $2.9\times10^{-3}$ | | Yaws (2003) | X | 237 |
| [53897-51-1] | $6.1\times10^{-4}$ | | Gharagheizi et al. (2012) | Q | |
| AUNQXXJGFDKEMS-UHFFFAOYSA-N | $3.2\times10^{-3}$ | | Gharagheizi et al. (2010) | Q | 246 |
| 3-ethyl-2-thiapentane | $3.0\times10^{-3}$ | | Yaws et al. (2003) | V | 802 |
| $C_6H_{14}S$ | $3.0\times10^{-3}$ | | Yaws (2003) | X | 237 |
| [57093-84-2] | $6.6\times10^{-4}$ | | Gharagheizi et al. (2012) | Q | |
| ZQOHJVSSVAWXQZ-UHFFFAOYSA-N | $2.8\times10^{-3}$ | | Gharagheizi et al. (2010) | Q | 246 |



Table A9.1: Sulfur (C, H, O, N, Cl, S) (... continued)

| Substance<br>Formula<br>(Trivial Name)<br>[CAS Registry Number]<br>InChIKey | $H_s^{cp}$<br>(at $T^{\ominus}$)<br>$\left[\dfrac{\mathrm{mol}}{\mathrm{m^3\,Pa}}\right]$ | $\dfrac{\mathrm{d}\ln H_s^{cp}}{\mathrm{d}(1/T)}$<br><br>[K] | Reference | Type | Note |
|---|---|---|---|---|---|
| 3-methyl-2-thiahexane<br>$C_6H_{14}S$<br>[13286-91-4]<br>WGBHWWSSUGCSCP-UHFFFAOYSA-N | $2.8\times10^{-3}$<br>$2.7\times10^{-3}$<br>$1.2\times10^{-3}$<br>$2.6\times10^{-3}$ | | Yaws et al. (2003)<br>Yaws (2003)<br>Gharagheizi et al. (2012)<br>Gharagheizi et al. (2010) | V<br>X<br>Q<br>Q | 802<br>237<br><br>246 |
| 4,4-dimethyl-2-thiapentane<br>$C_6H_{14}S$<br>[6079-57-8]<br>YUFAJXXIXSYTMS-UHFFFAOYSA-N | $2.7\times10^{-3}$<br>$2.7\times10^{-3}$<br>$3.1\times10^{-3}$ | | Yaws et al. (2003)<br>Yaws (2003)<br>Gharagheizi et al. (2010) | V<br>X<br>Q | 802<br>237<br>246 |
| 4-methyl-2-thiahexane<br>$C_6H_{14}S$<br>[15013-37-3]<br>XMOUDDXUCXXDTJ-UHFFFAOYSA-N | $2.7\times10^{-3}$<br>$2.7\times10^{-3}$<br>$3.9\times10^{-4}$<br>$2.9\times10^{-3}$ | | Yaws et al. (2003)<br>Yaws (2003)<br>Gharagheizi et al. (2012)<br>Gharagheizi et al. (2010) | V<br>X<br>Q<br>Q | 802<br>237<br><br>246 |
| 4-methyl-3-thiahexane<br>$C_6H_{14}S$<br>[5008-72-0]<br>JFNGZXUPUVUYST-UHFFFAOYSA-N | $2.7\times10^{-3}$<br>$2.7\times10^{-3}$<br>$2.2\times10^{-3}$<br>$2.6\times10^{-3}$ | | Yaws et al. (2003)<br>Yaws (2003)<br>Gharagheizi et al. (2012)<br>Gharagheizi et al. (2010) | V<br>X<br>Q<br>Q | 802<br>237<br><br>246 |
| 5-methyl-2-thiahexane<br>$C_6H_{14}S$<br>[13286-90-3]<br>ABIKQXWLJOURPN-UHFFFAOYSA-N | $2.8\times10^{-3}$<br>$2.8\times10^{-3}$<br>$5.4\times10^{-4}$<br>$2.7\times10^{-3}$ | | Yaws et al. (2003)<br>Yaws (2003)<br>Gharagheizi et al. (2012)<br>Gharagheizi et al. (2010) | V<br>X<br>Q<br>Q | 802<br>237<br><br>246 |
| 5-methyl-3-thiahexane<br>$C_6H_{14}S$<br>[1613-45-2]<br>OIRKGXWQBSPXLQ-UHFFFAOYSA-N | $2.7\times10^{-3}$<br>$2.7\times10^{-3}$<br>$7.1\times10^{-4}$<br>$2.5\times10^{-3}$ | | Yaws et al. (2003)<br>Yaws (2003)<br>Gharagheizi et al. (2012)<br>Gharagheizi et al. (2010) | V<br>X<br>Q<br>Q | 802<br>237<br><br>246 |
| ethyl butyl sulfide<br>$C_6H_{14}S$<br>[638-46-0]<br>XJIRSLHMKBUGMR-UHFFFAOYSA-N | $2.6\times10^{-3}$<br>$2.6\times10^{-3}$<br>$1.2\times10^{-3}$<br>$2.2\times10^{-3}$<br>$1.9\times10^{-3}$<br>$1.2\times10^{-3}$ | | Yaws et al. (2003)<br>Yaws (2003)<br>Gharagheizi et al. (2012)<br>Gharagheizi et al. (2010)<br>Yao et al. (2002)<br>Yaws (1999) | V<br>X<br>Q<br>Q<br>Q<br>? | 802<br>237<br><br>246<br>229<br>21 |
| methyl pentyl sulfide<br>$C_6H_{14}S$<br>[1741-83-9]<br>FOJGPFUFFHWGFQ-UHFFFAOYSA-N | $3.8\times10^{-3}$<br>$3.8\times10^{-3}$<br>$2.3\times10^{-3}$ | | Yaws et al. (2003)<br>Yaws (2003)<br>Gharagheizi et al. (2010) | V<br>X<br>Q | 802<br>237<br>246 |
| ethyl pentyl sulfide<br>$C_7H_{16}S$<br>[26158-99-6]<br>SOGIWVXLDPPMMF-UHFFFAOYSA-N | $1.0\times10^{-3}$<br>$9.6\times10^{-4}$<br>$1.1\times10^{-3}$<br>$1.8\times10^{-3}$ | | Yaws et al. (2003)<br>Yaws (2003)<br>Gharagheizi et al. (2012)<br>Gharagheizi et al. (2010) | V<br>X<br>Q<br>Q | 802<br>237<br><br>246 |
| methyl hexyl sulfide<br>$C_7H_{16}S$<br>[20291-60-5]<br>LZRXQHHKXDXOIC-UHFFFAOYSA-N | $1.0\times10^{-3}$<br>$9.6\times10^{-4}$<br>$5.2\times10^{-4}$<br>$1.8\times10^{-3}$ | | Yaws et al. (2003)<br>Yaws (2003)<br>Gharagheizi et al. (2012)<br>Gharagheizi et al. (2010) | V<br>X<br>Q<br>Q | 802<br>237<br><br>246 |



Table A9.1: Sulfur (C, H, O, N, Cl, S) (... continued)

| Substance Formula (Trivial Name) [CAS Registry Number] InChIKey | $H_s^{cp}$ (at $T^{\ominus}$) $\left[\dfrac{\mathrm{mol}}{\mathrm{m^3\,Pa}}\right]$ | $\dfrac{\mathrm{d\ln} H_s^{cp}}{\mathrm{d}(1/T)}$ [K] | Reference | Type | Note |
|---|---|---|---|---|---|
| propyl butyl sulfide | $1.0\times10^{-3}$ | | Yaws et al. (2003) | V | 802 |
| C$_7$H$_{16}$S | $9.6\times10^{-4}$ | | Yaws (2003) | X | 237 |
| [1613-46-3] | $7.6\times10^{-4}$ | | Gharagheizi et al. (2012) | Q | |
| ZBRWJPVULTZZCE-UHFFFAOYSA-N | $1.8\times10^{-3}$ | | Gharagheizi et al. (2010) | Q | 246 |
| dibutyl sulfide | $1.3\times10^{-3}$ | | Yaws et al. (2003) | V | 802 |
| C$_8$H$_{18}$S | $1.3\times10^{-3}$ | | Yaws (2003) | X | 237 |
| [544-40-1] | $3.9\times10^{-4}$ | | Gharagheizi et al. (2012) | Q | |
| HTIRHQRTDBPHNZ-UHFFFAOYSA-N | $1.5\times10^{-3}$ | | Gharagheizi et al. (2010) | Q | 246 |
| ethyl hexyl sulfide | $6.2\times10^{-4}$ | | Yaws et al. (2003) | V | 802 |
| C$_8$H$_{18}$S | $6.2\times10^{-4}$ | | Yaws (2003) | X | 237 |
| [7309-44-6] | $8.7\times10^{-4}$ | | Gharagheizi et al. (2012) | Q | |
| MGVUJBCOCITTRS-UHFFFAOYSA-N | $1.5\times10^{-3}$ | | Gharagheizi et al. (2010) | Q | 246 |
| methyl heptyl sulfide | $6.2\times10^{-4}$ | | Yaws et al. (2003) | V | 802 |
| C$_8$H$_{18}$S | $6.2\times10^{-4}$ | | Yaws (2003) | X | 237 |
| [20291-61-6] | $4.0\times10^{-4}$ | | Gharagheizi et al. (2012) | Q | |
| FJDWJOQOEZRIDJ-UHFFFAOYSA-N | $1.5\times10^{-3}$ | | Gharagheizi et al. (2010) | Q | 246 |
| propyl pentyl sulfide | $6.2\times10^{-4}$ | | Yaws et al. (2003) | V | 802 |
| C$_8$H$_{18}$S | $6.2\times10^{-4}$ | | Yaws (2003) | X | 237 |
| [42841-80-5] | $5.9\times10^{-4}$ | | Gharagheizi et al. (2012) | Q | |
| MJRCCWJSYFOGBX-UHFFFAOYSA-N | $1.5\times10^{-3}$ | | Gharagheizi et al. (2010) | Q | 246 |
| butyl pentyl sulfide | $9.8\times10^{-4}$ | | Yaws et al. (2003) | V | 802 |
| C$_9$H$_{20}$S | $9.7\times10^{-4}$ | | Yaws (2003) | X | 237 |
| [24768-42-1] | $7.0\times10^{-4}$ | | Gharagheizi et al. (2012) | Q | |
| RNEUXBDXTNIASG-UHFFFAOYSA-N | $1.4\times10^{-3}$ | | Gharagheizi et al. (2010) | Q | 246 |
| ethyl heptyl sulfide | $9.8\times10^{-4}$ | | Yaws et al. (2003) | V | 802 |
| C$_9$H$_{20}$S | $9.7\times10^{-4}$ | | Yaws (2003) | X | 237 |
| [24768-44-3] | $1.0\times10^{-3}$ | | Gharagheizi et al. (2012) | Q | |
| PYPULUCCVXMPFP-UHFFFAOYSA-N | $1.3\times10^{-3}$ | | Gharagheizi et al. (2010) | Q | 246 |
| methyl octyl sulfide | $9.8\times10^{-4}$ | | Yaws et al. (2003) | V | 802 |
| C$_9$H$_{20}$S | $9.7\times10^{-4}$ | | Yaws (2003) | X | 237 |
| [3698-95-1] | $4.7\times10^{-4}$ | | Gharagheizi et al. (2012) | Q | |
| AHCJTMBRROLNHV-UHFFFAOYSA-N | $1.3\times10^{-3}$ | | Gharagheizi et al. (2010) | Q | 246 |
| propyl hexyl sulfide | $9.8\times10^{-4}$ | | Yaws et al. (2003) | V | 802 |
| C$_9$H$_{20}$S | $9.7\times10^{-4}$ | | Yaws (2003) | X | 237 |
| [24768-43-2] | $7.0\times10^{-4}$ | | Gharagheizi et al. (2012) | Q | |
| ABZLKKGJOVPBBL-UHFFFAOYSA-N | $1.4\times10^{-3}$ | | Gharagheizi et al. (2010) | Q | 246 |
| butyl hexyl sulfide | $1.6\times10^{-3}$ | | Yaws et al. (2003) | V | 802 |
| C$_{10}$H$_{22}$S | $1.6\times10^{-3}$ | | Yaws (2003) | X | 237 |
| [16967-04-7] | $8.1\times10^{-4}$ | | Gharagheizi et al. (2012) | Q | |
| YZUHMAFUXBPUKH-UHFFFAOYSA-N | $1.3\times10^{-3}$ | | Gharagheizi et al. (2010) | Q | 246 |





Table A9.1: Sulfur (C, H, O, N, Cl, S) (. . . continued)

| Substance<br>Formula<br>(Trivial Name)<br>[CAS Registry Number]<br>InChIKey | $H_s^{cp}$<br>(at $T^{\ominus}$)<br>$\left[\dfrac{\mathrm{mol}}{\mathrm{m}^3\,\mathrm{Pa}}\right]$ | $\dfrac{\mathrm{d}\ln H_s^{cp}}{\mathrm{d}(1/T)}$<br><br>[K] | Reference | Type | Note |
|---|---|---|---|---|---|
| dipentyl sulfide | $1.6\times10^{-3}$ | | Yaws et al. (2003) | V | 802 |
| $C_{10}H_{22}S$ | $1.6\times10^{-3}$ | | Yaws (2003) | X | 237 |
| [872-10-6] | $8.1\times10^{-4}$ | | Gharagheizi et al. (2012) | Q | |
| JOZDADPMWLVEJK-UHFFFAOYSA-N | $1.3\times10^{-3}$ | | Gharagheizi et al. (2010) | Q | 246 |
| ethyl octyl sulfide | $1.6\times10^{-3}$ | | Yaws et al. (2003) | V | 802 |
| $C_{10}H_{22}S$ | $1.6\times10^{-3}$ | | Yaws (2003) | X | 237 |
| [3698-94-0] | $1.2\times10^{-3}$ | | Gharagheizi et al. (2012) | Q | |
| WAITXWGCJQLPGH-UHFFFAOYSA-N | $1.3\times10^{-3}$ | | Gharagheizi et al. (2010) | Q | 246 |
| methyl nonyl sulfide | $1.6\times10^{-3}$ | | Yaws et al. (2003) | V | 802 |
| $C_{10}H_{22}S$ | $1.6\times10^{-3}$ | | Yaws (2003) | X | 237 |
| [59973-07-8] | $3.6\times10^{-4}$ | | Gharagheizi et al. (2012) | Q | |
| FCRSULZJMFDBIK-UHFFFAOYSA-N | $1.3\times10^{-3}$ | | Gharagheizi et al. (2010) | Q | 246 |
| propyl heptyl sulfide | $1.6\times10^{-3}$ | | Yaws et al. (2003) | V | 802 |
| $C_{10}H_{22}S$ | $1.6\times10^{-3}$ | | Yaws (2003) | X | 237 |
| [24768-46-5] | $8.1\times10^{-4}$ | | Gharagheizi et al. (2012) | Q | |
| PCPVCHFKVPNTBH-UHFFFAOYSA-N | $1.3\times10^{-3}$ | | Gharagheizi et al. (2010) | Q | 246 |
| butyl heptyl sulfide | $1.7\times10^{-3}$ | | Yaws et al. (2003) | V | 802 |
| $C_{11}H_{24}S$ | $1.7\times10^{-3}$ | | Yaws (2003) | X | 237 |
| [40813-84-1] | $9.1\times10^{-4}$ | | Gharagheizi et al. (2012) | Q | |
| HYUPOCGXBZUYFY-UHFFFAOYSA-N | $1.4\times10^{-3}$ | | Gharagheizi et al. (2010) | Q | 246 |
| ethyl nonyl sulfide | $1.7\times10^{-3}$ | | Yaws et al. (2003) | V | 802 |
| $C_{11}H_{24}S$ | $1.7\times10^{-3}$ | | Yaws (2003) | X | 237 |
| [59973-08-9] | $8.8\times10^{-4}$ | | Gharagheizi et al. (2012) | Q | |
| LUAABLIRQSWMGZ-UHFFFAOYSA-N | $1.4\times10^{-3}$ | | Gharagheizi et al. (2010) | Q | 246 |
| methyl decyl sulfide | $1.7\times10^{-3}$ | | Yaws et al. (2003) | V | 802 |
| $C_{11}H_{24}S$ | $1.7\times10^{-3}$ | | Yaws (2003) | X | 237 |
| [22438-39-7] | $4.1\times10^{-4}$ | | Gharagheizi et al. (2012) | Q | |
| HKGUUZAACYBIID-UHFFFAOYSA-N | $1.4\times10^{-3}$ | | Gharagheizi et al. (2010) | Q | 246 |
| propyl octyl sulfide | $1.7\times10^{-3}$ | | Yaws et al. (2003) | V | 802 |
| $C_{11}H_{24}S$ | $1.7\times10^{-3}$ | | Yaws (2003) | X | 237 |
| [3698-93-9] | $9.1\times10^{-4}$ | | Gharagheizi et al. (2012) | Q | |
| GPJXDRJGQAKGLH-UHFFFAOYSA-N | $1.4\times10^{-3}$ | | Gharagheizi et al. (2010) | Q | 246 |
| butyl octyl sulfide | $1.6\times10^{-3}$ | | Yaws et al. (2003) | V | 802 |
| $C_{12}H_{26}S$ | $1.5\times10^{-3}$ | | Yaws (2003) | X | 237 |
| [16900-07-5] | $1.0\times10^{-3}$ | | Gharagheizi et al. (2012) | Q | |
| UNIAPWPIAGJFDG-UHFFFAOYSA-N | $1.8\times10^{-3}$ | | Gharagheizi et al. (2010) | Q | 246 |
| dihexyl sulfide | $1.6\times10^{-3}$ | | Yaws et al. (2003) | V | 802 |
| $C_{12}H_{26}S$ | $1.5\times10^{-3}$ | | Yaws (2003) | X | 237 |
| [6294-31-1] | $1.0\times10^{-3}$ | | Gharagheizi et al. (2012) | Q | |
| LHNRHYOMDUJLLM-UHFFFAOYSA-N | $1.8\times10^{-3}$ | | Gharagheizi et al. (2010) | Q | 246 |



Table A9.1: Sulfur (C, H, O, N, Cl, S) (...continued)

| Substance<br>Formula<br>(Trivial Name)<br>[CAS Registry Number]<br>InChIKey | $H_s^{cp}$<br>(at $T^\ominus$)<br>$\left[\dfrac{\mathrm{mol}}{\mathrm{m^3\,Pa}}\right]$ | $\dfrac{\mathrm{d}\ln H_s^{cp}}{\mathrm{d}(1/T)}$<br><br>[K] | Reference | Type | Note |
|---|---|---|---|---|---|
| ethyl decyl sulfide | $1.6\times10^{-3}$ | | Yaws et al. (2003) | V | 802 |
| $C_{12}H_{26}S$ | $1.5\times10^{-3}$ | | Yaws (2003) | X | 237 |
| [19313-61-2] | $9.7\times10^{-4}$ | | Gharagheizi et al. (2012) | Q | |
| VSSRSPLEFYQIEK-UHFFFAOYSA-N | $1.8\times10^{-3}$ | | Gharagheizi et al. (2010) | Q | 246 |
| methyl undecyl sulfide | $1.6\times10^{-3}$ | | Yaws et al. (2003) | V | 802 |
| $C_{12}H_{26}S$ | $1.5\times10^{-3}$ | | Yaws (2003) | X | 237 |
| [7289-44-3] | $4.5\times10^{-4}$ | | Gharagheizi et al. (2012) | Q | |
| HDOADYQJIBYVGE-UHFFFAOYSA-N | $1.7\times10^{-3}$ | | Gharagheizi et al. (2010) | Q | 246 |
| propyl nonyl sulfide | $1.6\times10^{-3}$ | | Yaws et al. (2003) | V | 802 |
| $C_{12}H_{26}S$ | $1.5\times10^{-3}$ | | Yaws (2003) | X | 237 |
| [62103-66-6] | $6.6\times10^{-4}$ | | Gharagheizi et al. (2012) | Q | |
| AQAOPNMAEGBYHI-UHFFFAOYSA-N | $1.8\times10^{-3}$ | | Gharagheizi et al. (2010) | Q | 246 |
| butyl nonyl sulfide | $1.8\times10^{-3}$ | | Yaws et al. (2003) | V | 802 |
| $C_{13}H_{28}S$ | $1.8\times10^{-3}$ | | Yaws (2003) | X | 237 |
| [66577-32-0] | $7.1\times10^{-4}$ | | Gharagheizi et al. (2012) | Q | |
| FWRIVMHSSSZAFD-UHFFFAOYSA-N | $2.6\times10^{-3}$ | | Gharagheizi et al. (2010) | Q | 246 |
| ethyl undecyl sulfide | $1.8\times10^{-3}$ | | Yaws et al. (2003) | V | 802 |
| $C_{13}H_{28}S$ | $1.8\times10^{-3}$ | | Yaws (2003) | X | 237 |
| [66577-30-8] | $1.0\times10^{-3}$ | | Gharagheizi et al. (2012) | Q | |
| OSWITQLVZPPUIR-UHFFFAOYSA-N | $2.5\times10^{-3}$ | | Gharagheizi et al. (2010) | Q | 246 |
| methyl dodecyl sulfide | $1.8\times10^{-3}$ | | Yaws et al. (2003) | V | 802 |
| $C_{13}H_{28}S$ | $1.8\times10^{-3}$ | | Yaws (2003) | X | 237 |
| [3698-89-3] | $4.8\times10^{-4}$ | | Gharagheizi et al. (2012) | Q | |
| KJWHJDGMOQJLGF-UHFFFAOYSA-N | $2.5\times10^{-3}$ | | Gharagheizi et al. (2010) | Q | 246 |
| propyl decyl sulfide | $1.8\times10^{-3}$ | | Yaws et al. (2003) | V | 802 |
| $C_{13}H_{28}S$ | $1.8\times10^{-3}$ | | Yaws (2003) | X | 237 |
| [66577-31-9] | $7.1\times10^{-4}$ | | Gharagheizi et al. (2012) | Q | |
| HPJCKXDELORROO-UHFFFAOYSA-N | $2.6\times10^{-3}$ | | Gharagheizi et al. (2010) | Q | 246 |
| butyl decyl sulfide | $2.6\times10^{-3}$ | | Yaws et al. (2003) | V | 802 |
| $C_{14}H_{30}S$ | $2.6\times10^{-3}$ | | Yaws (2003) | X | 237 |
| [19313-57-6] | $7.5\times10^{-4}$ | | Gharagheizi et al. (2012) | Q | |
| CLBLVLKZMJDLOT-UHFFFAOYSA-N | $4.7\times10^{-3}$ | | Gharagheizi et al. (2010) | Q | 246 |
| diheptyl sulfide | $2.6\times10^{-3}$ | | Yaws et al. (2003) | V | 802 |
| $C_{14}H_{30}S$ | $2.6\times10^{-3}$ | | Yaws (2003) | X | 237 |
| [629-65-2] | $1.2\times10^{-3}$ | | Gharagheizi et al. (2012) | Q | |
| LEMIDOZYVQXGLI-UHFFFAOYSA-N | $4.7\times10^{-3}$ | | Gharagheizi et al. (2010) | Q | 246 |
| ethyl dodecyl sulfide | $2.6\times10^{-3}$ | | Yaws et al. (2003) | V | 802 |
| $C_{14}H_{30}S$ | $2.6\times10^{-3}$ | | Yaws (2003) | X | 237 |
| [2851-83-4] | $1.1\times10^{-3}$ | | Gharagheizi et al. (2012) | Q | |
| QECBTJWQRXCSCU-UHFFFAOYSA-N | $4.4\times10^{-3}$ | | Gharagheizi et al. (2010) | Q | 246 |



Table A9.1: Sulfur (C, H, O, N, Cl, S) (... continued)

| Substance Formula (Trivial Name) [CAS Registry Number] InChIKey | $H_s^{cp}$ (at $T^\ominus$) $\left[\dfrac{\mathrm{mol}}{\mathrm{m^3\,Pa}}\right]$ | $\dfrac{\mathrm{d}\ln H_s^{cp}}{\mathrm{d}(1/T)}$ [K] | Reference | Type | Note |
|---|---|---|---|---|---|
| methyl tridecyl sulfide | $2.6\times10^{-3}$ | | Yaws et al. (2003) | V | 802 |
| $C_{14}H_{30}S$ | $2.6\times10^{-3}$ | | Yaws (2003) | X | 237 |
| [62155-09-3] | $5.1\times10^{-4}$ | | Gharagheizi et al. (2012) | Q | |
| QVXOOYOFFITGCV-UHFFFAOYSA-N | $4.5\times10^{-3}$ | | Gharagheizi et al. (2010) | Q | 246 |
| propyl undecyl sulfide | $2.6\times10^{-3}$ | | Yaws et al. (2003) | V | 802 |
| $C_{14}H_{30}S$ | $2.6\times10^{-3}$ | | Yaws (2003) | X | 237 |
| [66826-84-4] | $7.5\times10^{-4}$ | | Gharagheizi et al. (2012) | Q | |
| ITNOZFAANYXVSG-UHFFFAOYSA-N | $4.7\times10^{-3}$ | | Gharagheizi et al. (2010) | Q | 246 |
| dimethyl disulfide | $5.8\times10^{-3}$ | | Burkholder et al. (2019) | L | |
| $CH_3SSCH_3$ | $5.8\times10^{-3}$ | | Burkholder et al. (2015) | L | |
| [624-92-0] | $8.3\times10^{-3}$ | 3700 | Brockbank (2013) | L | 1 |
| WQOXQRCZOLPYPM-UHFFFAOYSA-N | $7.4\times10^{-3}$ | 4200 | Plyasunova et al. (2004) | L | |
| | $6.7\times10^{-3}$ | 5200 | Bruneel et al. (2016) | M | |
| | $5.8\times10^{-3}$ | | Schuhfried et al. (2011) | M | |
| | $6.5\times10^{-3}$ | 3200 | Falabella (2007) | M | 11, 338 |
| | $9.1\times10^{-3}$ | 4100 | Iliuta and Larachi (2005b) | M | |
| | $7.8\times10^{-3}$ | | Souchon et al. (2004) | M | |
| | $5.9\times10^{-3}$ | | Pollien et al. (2003) | M | |
| | $3.6\times10^{-3}$ | | McIntosh and Heffron (2000) | M | 14 |
| | $9.4\times10^{-3}$ | 4300 | Przyjazny et al. (1983) | M | |
| | $8.6\times10^{-3}$ | | Mazza (1980) | M | |
| | $8.3\times10^{-3}$ | | Vitenberg et al. (1975) | M | 12 |
| | $1.7\times10^{-2}$ | | Mackay et al. (2006d) | V | |
| | $1.7\times10^{-2}$ | | Mackay et al. (1995) | V | |
| | $9.0\times10^{-3}$ | | Vitenberg et al. (1975) | R | 12 |
| | $3.0\times10^{-2}$ | | Hilal et al. (2008) | Q | |
| | $8.7\times10^{-2}$ | | Modarresi et al. (2007) | Q | 67 |
| | | 1700 | Kühne et al. (2005) | Q | |
| | $4.6\times10^{-3}$ | | Nirmalakhandan et al. (1997) | Q | |
| | | 1600 | Kühne et al. (2005) | ? | |
| | $9.0\times10^{-3}$ | | Abraham et al. (1990) | ? | |
| diethyl disulfide | $3.7\times10^{-3}$ | | Burkholder et al. (2019) | L | |
| $C_2H_5SSC_2H_5$ | $3.7\times10^{-3}$ | | Burkholder et al. (2015) | L | |
| [110-81-6] | $4.0\times10^{-3}$ | 4900 | Plyasunova et al. (2004) | L | |
| CETBSQOFQKLHHZ-UHFFFAOYSA-N | $3.7\times10^{-3}$ | | Schuhfried et al. (2011) | M | |
| | $6.3\times10^{-3}$ | 4300 | Przyjazny et al. (1983) | M | |
| | $4.7\times10^{-3}$ | | Vitenberg et al. (1975) | M | 12 |
| | $1.2\times10^{-2}$ | | Hilal et al. (2008) | Q | |
| | $3.7\times10^{-2}$ | | Modarresi et al. (2007) | Q | 67 |
| | $2.3\times10^{-3}$ | | Nirmalakhandan et al. (1997) | Q | |
| | $6.4\times10^{-3}$ | | Abraham et al. (1990) | ? | |
| dipropyl disulfide | $2.2\times10^{-3}$ | 5400 | Plyasunova et al. (2004) | L | |
| $C_3H_7SSC_3H_7$ | $2.4\times10^{-3}$ | | Schuhfried et al. (2011) | M | |
| [629-19-6] | $1.8\times10^{-2}$ | | Mazza (1980) | M | |
| ALVPFGSHPUPROW-UHFFFAOYSA-N | | | | | |



Table A9.1: Sulfur (C, H, O, N, Cl, S) (. . . continued)

| Substance<br>Formula<br>(Trivial Name)<br>[CAS Registry Number]<br>InChIKey | $H_s^{cp}$<br>(at $T^{\ominus}$)<br>$\left[\dfrac{\text{mol}}{\text{m}^3\,\text{Pa}}\right]$ | $\dfrac{\text{d}\ln H_s^{cp}}{\text{d}(1/T)}$<br><br>[K] | Reference | Type | Note |
|---|---|---|---|---|---|
| carbon disulfide | $6.1\times10^{-4}$ | 4300 | Burkholder et al. (2019) | L | |
| $CS_2$ | $6.1\times10^{-4}$ | 4300 | Burkholder et al. (2015) | L | |
| [75-15-0] | $6.1\times10^{-4}$ | 3900 | Warneck and Williams (2012) | L | |
| QGJOPFRUJISHPQ-UHFFFAOYSA-N | $6.1\times10^{-4}$ | 4300 | Sander et al. (2011) | L | |
| | $6.1\times10^{-4}$ | 4300 | Sander et al. (2006) | L | |
| | $5.5\times10^{-4}$ | 3700 | Plyasunova et al. (2004) | L | |
| | $5.7\times10^{-4}$ | 3800 | Hiatt (2013) | M | |
| | $5.4\times10^{-4}$ | 2800 | De Bruyn et al. (1995b) | M | |
| | $6.2\times10^{-4}$ | 3800 | Elliott (1989) | M | |
| | $5.4\times10^{-4}$ | 4300 | Rex (1906) | M | |
| | $5.7\times10^{-4}$ | | Mackay et al. (2006d) | V | |
| | $5.7\times10^{-4}$ | | Mackay et al. (1995) | V | |
| | $8.0\times10^{-4}$ | | Hwang et al. (1992) | V | |
| | $4.5\times10^{-4}$ | 4100 | Winkler (1906) | V | |
| | $5.1\times10^{-4}$ | | Yaws (2003) | X | 237 |
| | $7.5\times10^{-4}$ | 1200 | Goldstein (1982) | X | 298 |
| | $5.0\times10^{-4}$ | | Hayer et al. (2022) | Q | 20 |
| | $1.3\times10^{-2}$ | | Gharagheizi et al. (2012) | Q | |
| | $5.1\times10^{-4}$ | | Gharagheizi et al. (2010) | Q | 246 |
| | $9.4\times10^{-5}$ | | Yaws (1999) | ? | 21 |
| | $3.2\times10^{-4}$ | | Abraham and Weathersby (1994) | ? | 21 |
| | $5.1\times10^{-4}$ | | Yaws and Yang (1992) | ? | 21 |
| | | | Schäfer and Lax (1962) | ? | 808 |
| | | | Booth and Jolley (1943) | ? | 809 |
| | | | Booth and Jolley (1943) | ? | 810 |
| 2,3,4-trithiapentane | $1.2\times10^{-2}$ | | Plyasunova et al. (2004) | L | |
| $C_2H_6S_3$ | $1.4\times10^{-2}$ | | Souchon et al. (2004) | M | |
| (dimethyl trisulfide) | $2.1\times10^{-2}$ | | Roberts and Pollien (1997) | M | |
| [3658-80-8] | | | | | |
| YWHLKYXPLRWGSE-UHFFFAOYSA-N | | | | | |
| dicyclohexyldisulfide | $2.5\times10^{-3}$ | | HSDB (2015) | Q | 99 |
| $C_{12}H_{22}S_2$ | | | | | |
| [2550-40-5] | | | | | |
| ODHAQPXNQDBHSH-UHFFFAOYSA-N | | | | | |
| allyl mercaptan | $1.2\times10^{-2}$ | | Hilal et al. (2008) | Q | |
| $C_3H_6S$ | | | | | |
| [870-23-5] | | | | | |
| ULIKDJVNUXNQHS-UHFFFAOYSA-N | | | | | |
| 3,3'-thiobis-1-propene | $4.1\times10^{-3}$ | | Lindinger et al. (1998) | M | 811 |
| $(C_3H_5)_2S$ | $7.1\times10^{-3}$ | | Mazza (1980) | M | |
| (diallyl sulfide) | $7.6\times10^{-3}$ | | HSDB (2015) | Q | 99 |
| [592-88-1] | $9.9\times10^{-3}$ | | Hilal et al. (2008) | Q | |
| UBJVUCKUDDKUJF-UHFFFAOYSA-N | | | | | |





Table A9.1: Sulfur (C, H, O, N, Cl, S) (...continued)

| Substance Formula (Trivial Name) [CAS Registry Number] InChIKey | $H_s^{cp}$ (at $T^{\ominus}$) $\left[\dfrac{\text{mol}}{\text{m}^3\,\text{Pa}}\right]$ | $\dfrac{\mathrm{d}\ln H_s^{cp}}{\mathrm{d}(1/T)}$ [K] | Reference | Type | Note |
|---|---|---|---|---|---|
| allyl methyl sulfide | $4.2\times10^{-3}$ | | Burkholder et al. (2019) | L | |
| $CH_2CHCH_2SCH_3$ | $4.2\times10^{-3}$ | | Burkholder et al. (2015) | L | |
| (2-propenyl methyl sulfide) | $4.2\times10^{-3}$ | | Schuhfried et al. (2011) | M | |
| [10152-76-8] | $5.0\times10^{-3}$ | | Mazza (1980) | M | |
| NVLPQIPTCCLBEU-UHFFFAOYSA-N | | | | | |
| thiophene | $4.1\times10^{-3}$ | 4300 | Haimi et al. (2006) | M | 812 |
| $C_4H_4S$ | $4.4\times10^{-3}$ | 4000 | Przyjazny et al. (1983) | M | |
| [110-02-1] | $3.4\times10^{-3}$ | | HSDB (2015) | V | |
| YTPLMLYBLZKORZ-UHFFFAOYSA-N | | | Mackay et al. (2006d) | V | 558 |
| | $4.5\times10^{-3}$ | | Mackay et al. (1995) | V | |
| | $3.4\times10^{-3}$ | | Yaws (2003) | X | 237 |
| | $8.2\times10^{-3}$ | | Gharagheizi et al. (2012) | Q | |
| | $3.1\times10^{-3}$ | | Gharagheizi et al. (2010) | Q | 246 |
| | $1.4\times10^{-3}$ | | Hilal et al. (2008) | Q | |
| | $9.8\times10^{-3}$ | | Modarresi et al. (2007) | Q | 67 |
| | | 2800 | Kühne et al. (2005) | Q | |
| | $3.4\times10^{-3}$ | | Yaffe et al. (2003) | Q | 248, 249 |
| | $9.3\times10^{-3}$ | | Yao et al. (2002) | Q | 229, 267 |
| | $8.8\times10^{-3}$ | | English and Carroll (2001) | Q | 230, 231 |
| | $6.9\times10^{-4}$ | | Katritzky et al. (1998) | Q | |
| | $4.5\times10^{-3}$ | | Mackay et al. (2006d) | ? | |
| | | 1900 | Kühne et al. (2005) | ? | |
| | $3.4\times10^{-3}$ | | Yaws et al. (2003) | ? | 21 |
| | $3.4\times10^{-3}$ | | Yaws (1999) | ? | 21 |
| | $3.4\times10^{-3}$ | | Yaws and Yang (1992) | ? | 21 |
| | $4.4\times10^{-3}$ | | Abraham et al. (1990) | ? | |
| 2-methylthiophene | $4.1\times10^{-3}$ | 4300 | Brockbank (2013) | L | 1 |
| $CH_3C_4H_3S$ | $4.1\times10^{-3}$ | 4300 | Przyjazny et al. (1983) | M | |
| [554-14-3] | $2.2\times10^{-3}$ | | Yaws et al. (2003) | V | 802 |
| XQQBUAPQHNYYRS-UHFFFAOYSA-N | $1.9\times10^{-3}$ | | Yaws (2003) | X | 237 |
| | $2.9\times10^{-2}$ | | Keshavarz et al. (2022) | Q | |
| | $2.7\times10^{-2}$ | | Duchowicz et al. (2020) | Q | |
| | $1.1\times10^{-2}$ | | Gharagheizi et al. (2012) | Q | |
| | $1.9\times10^{-3}$ | | Gharagheizi et al. (2010) | Q | 246 |
| | $1.4\times10^{-3}$ | | Hilal et al. (2008) | Q | |
| | $8.0\times10^{-3}$ | | English and Carroll (2001) | Q | 230, 231 |
| | $4.1\times10^{-3}$ | | Duchowicz et al. (2020) | ? | 185, 21 |
| | $4.1\times10^{-3}$ | | Abraham et al. (1990) | ? | |
| 3-methylthiophene | $1.4\times10^{-3}$ | | Duchowicz et al. (2020) | V | 186 |
| $CH_3C_4H_3S$ | $1.4\times10^{-3}$ | | Yaws (2003) | X | 237 |
| [616-44-4] | $2.7\times10^{-2}$ | | Duchowicz et al. (2020) | Q | |
| QENGPZGAWFQWCZ-UHFFFAOYSA-N | $4.0\times10^{-3}$ | | Gharagheizi et al. (2012) | Q | |
| | $1.9\times10^{-3}$ | | Gharagheizi et al. (2010) | Q | 246 |
| | $1.7\times10^{-3}$ | | Hilal et al. (2008) | Q | |
| | $8.2\times10^{-3}$ | | Modarresi et al. (2007) | Q | 67 |
| | $1.4\times10^{-3}$ | | Yaffe et al. (2003) | Q | 248, 249 |





Table A9.1: Sulfur (C, H, O, N, Cl, S) (...continued)

| Substance Formula (Trivial Name) [CAS Registry Number] InChIKey | $H_s^{cp}$ (at $T^{\ominus}$) $\left[\dfrac{\text{mol}}{\text{m}^3\,\text{Pa}}\right]$ | $\dfrac{\text{d}\ln H_s^{cp}}{\text{d}(1/T)}$ [K] | Reference | Type | Note |
|---|---|---|---|---|---|
| | $1.8\times10^{-3}$ | | Katritzky et al. (1998) | Q | |
| | $1.4\times10^{-3}$ | | Yaws et al. (2003) | ? | 21 |
| propyl allyl disulfide $C_6H_{12}S_2$ [2179-59-1] FCSSPCOFDUKHPV-UHFFFAOYSA-N | $3.5\times10^{-3}$ | | HSDB (2015) | Q | 99 |
| 4,5-dithia-1,7-octadiene $C_6H_{10}S_2$ (diallyl disulfide) [2179-57-9] PFRGXCVKLLPLIP-UHFFFAOYSA-N | $7.5\times10^{-3}$ | | Mazza (1980) | M | |
| 2,3-dimethylthiophene $C_6H_8S$ [632-16-6] BZYUMXXOAYSFOW-UHFFFAOYSA-N | $1.5\times10^{-3}$ $1.3\times10^{-3}$ $4.8\times10^{-3}$ $1.4\times10^{-3}$ | | Yaws et al. (2003) Yaws (2003) Gharagheizi et al. (2012) Gharagheizi et al. (2010) | V X Q Q | 802 237 246 |
| 2,4-dimethylthiophene $C_6H_8S$ [638-00-6] CPULIKNSOUFMPL-UHFFFAOYSA-N | $1.5\times10^{-3}$ $1.3\times10^{-3}$ $4.7\times10^{-3}$ $1.4\times10^{-3}$ | | Yaws et al. (2003) Yaws (2003) Gharagheizi et al. (2012) Gharagheizi et al. (2010) | V X Q Q | 802 237 246 |
| 2,5-dimethylthiophene $C_6H_8S$ [638-02-8] GWQOOADXMVQEFT-UHFFFAOYSA-N | $3.7\times10^{-3}$ $1.6\times10^{-3}$ $1.4\times10^{-3}$ $1.3\times10^{-2}$ $1.4\times10^{-3}$ | | Mazza (1980) Yaws et al. (2003) Yaws (2003) Gharagheizi et al. (2012) Gharagheizi et al. (2010) | M V X Q Q | 802 237 246 |
| 2-ethylthiophene $C_6H_8S$ [872-55-9] JCCCMAAJYSNBPR-UHFFFAOYSA-N | $1.4\times10^{-3}$ $3.9\times10^{-3}$ $1.5\times10^{-3}$ $1.9\times10^{-3}$ | | Yaws (2003) Gharagheizi et al. (2012) Gharagheizi et al. (2010) Yaws et al. (2003) | X Q Q ? | 237 246 21 |
| 3,4-dimethylthiophene $C_6H_8S$ [632-15-5] GPSFYJDZKSRMKZ-UHFFFAOYSA-N | $1.5\times10^{-3}$ $1.2\times10^{-3}$ $1.7\times10^{-3}$ $1.4\times10^{-3}$ | | Yaws et al. (2003) Yaws (2003) Gharagheizi et al. (2012) Gharagheizi et al. (2010) | V X Q Q | 802 237 246 |
| 3-ethylthiophene $C_6H_8S$ [1795-01-3] SLDBAXYJAIRQMX-UHFFFAOYSA-N | $1.6\times10^{-3}$ $1.4\times10^{-3}$ $1.9\times10^{-3}$ $1.4\times10^{-3}$ | | Yaws et al. (2003) Yaws (2003) Gharagheizi et al. (2012) Gharagheizi et al. (2010) | V X Q Q | 802 237 246 |
| 2,3,4-trimethylthiophene $C_7H_{10}S$ [1795-04-6] MAVVDCDMBKFUES-UHFFFAOYSA-N | $1.3\times10^{-3}$ $1.1\times10^{-3}$ $2.9\times10^{-3}$ $1.0\times10^{-3}$ | | Yaws et al. (2003) Yaws (2003) Gharagheizi et al. (2012) Gharagheizi et al. (2010) | V X Q Q | 802 237 246 |





Table A9.1: Sulfur (C, H, O, N, Cl, S) (...continued)

| Substance Formula (Trivial Name) [CAS Registry Number] InChIKey | $H_s^{cp}$ (at $T^{\ominus}$) $\left[\dfrac{\text{mol}}{\text{m}^3\,\text{Pa}}\right]$ | $\dfrac{\text{d}\ln H_s^{cp}}{\text{d}(1/T)}$ [K] | Reference | Type | Note |
|---|---|---|---|---|---|
| 2,3,5-trimethylthiophene C$_7$H$_{10}$S [1795-05-7] QKZJQIHBRCFDGQ-UHFFFAOYSA-N | $1.1\times10^{-3}$ $9.3\times10^{-4}$ $7.2\times10^{-3}$ $9.7\times10^{-4}$ | | Yaws et al. (2003) Yaws (2003) Gharagheizi et al. (2012) Gharagheizi et al. (2010) | V X Q Q | 802 237 246 |
| 2-isopropylthiophene C$_7$H$_{10}$S [4095-22-1] LOXBELRNKUFSRD-UHFFFAOYSA-N | $1.3\times10^{-3}$ $1.1\times10^{-3}$ $1.8\times10^{-3}$ $1.1\times10^{-3}$ | | Yaws et al. (2003) Yaws (2003) Gharagheizi et al. (2012) Gharagheizi et al. (2010) | V X Q Q | 802 237 246 |
| 2-methyl-3-ethylthiophene C$_7$H$_{10}$S [53119-51-0] RBRAJDCWXUJHIY-UHFFFAOYSA-N | $1.2\times10^{-3}$ $1.0\times10^{-3}$ $2.8\times10^{-3}$ $1.0\times10^{-3}$ | | Yaws et al. (2003) Yaws (2003) Gharagheizi et al. (2012) Gharagheizi et al. (2010) | V X Q Q | 802 237 246 |
| 2-methyl-4-ethylthiophene C$_7$H$_{10}$S [13678-54-1] KIWVMUQUYOKTKU-UHFFFAOYSA-N | $1.1\times10^{-3}$ $9.6\times10^{-4}$ $3.3\times10^{-3}$ $1.0\times10^{-3}$ | | Yaws et al. (2003) Yaws (2003) Gharagheizi et al. (2012) Gharagheizi et al. (2010) | V X Q Q | 802 237 246 |
| 2-methyl-5-ethylthiophene C$_7$H$_{10}$S [40323-88-4] VOIVNYVBGCJFRW-UHFFFAOYSA-N | $1.2\times10^{-3}$ $9.9\times10^{-4}$ $6.6\times10^{-3}$ $1.1\times10^{-3}$ | | Yaws et al. (2003) Yaws (2003) Gharagheizi et al. (2012) Gharagheizi et al. (2010) | V X Q Q | 802 237 246 |
| 2-propylthiophene C$_7$H$_{10}$S [1551-27-5] BTXIJTYYMLCUHI-UHFFFAOYSA-N | $1.2\times10^{-3}$ $1.0\times10^{-3}$ $3.1\times10^{-3}$ $1.0\times10^{-3}$ | | Yaws et al. (2003) Yaws (2003) Gharagheizi et al. (2012) Gharagheizi et al. (2010) | V X Q Q | 802 237 246 |
| 3-isopropylthiophene C$_7$H$_{10}$S [29488-27-5] LJPDBPCGTFTUDE-UHFFFAOYSA-N | $1.1\times10^{-3}$ $9.7\times10^{-4}$ $1.4\times10^{-3}$ $1.0\times10^{-3}$ | | Yaws et al. (2003) Yaws (2003) Gharagheizi et al. (2012) Gharagheizi et al. (2010) | V X Q Q | 802 237 246 |
| 3-methyl-2-ethylthiophene C$_7$H$_{10}$S [31805-48-8] VVSCPYCHXFEKLF-UHFFFAOYSA-N | $1.2\times10^{-3}$ $9.8\times10^{-4}$ $2.1\times10^{-3}$ $1.1\times10^{-3}$ | | Yaws et al. (2003) Yaws (2003) Gharagheizi et al. (2012) Gharagheizi et al. (2010) | V X Q Q | 802 237 246 |
| 3-methyl-4-ethylthiophene C$_7$H$_{10}$S [66577-03-5] QMHNYDJBWOPGHE-UHFFFAOYSA-N | $1.3\times10^{-3}$ $1.1\times10^{-3}$ $8.1\times10^{-4}$ $1.0\times10^{-3}$ | | Yaws et al. (2003) Yaws (2003) Gharagheizi et al. (2012) Gharagheizi et al. (2010) | V X Q Q | 802 237 246 |
| 3-methyl-5-ethylthiophene C$_7$H$_{10}$S (2-ethyl-4-methylthiophene) [66577-04-6] NZOYEHPXDWJOCD-UHFFFAOYSA-N | $1.3\times10^{-3}$ $1.1\times10^{-3}$ $1.8\times10^{-3}$ $1.1\times10^{-3}$ | | Yaws et al. (2003) Yaws (2003) Gharagheizi et al. (2012) Gharagheizi et al. (2010) | V X Q Q | 802 237 246 |





Table A9.1: Sulfur (C, H, O, N, Cl, S) (. . . continued)

| Substance Formula (Trivial Name) [CAS Registry Number] InChIKey | $H_s^{cp}$ (at $T^{\ominus}$) $\left[\dfrac{\mathrm{mol}}{\mathrm{m}^3\,\mathrm{Pa}}\right]$ | $\dfrac{\mathrm{d}\ln H_s^{cp}}{\mathrm{d}(1/T)}$ [K] | Reference | Type | Note |
|---|---|---|---|---|---|
| 3-propylthiophene | $1.2\times10^{-3}$ | | Yaws et al. (2003) | V | 802 |
| C$_7$H$_{10}$S | $9.9\times10^{-4}$ | | Yaws (2003) | X | 237 |
| [1518-75-8] | $1.5\times10^{-3}$ | | Gharagheizi et al. (2012) | Q | |
| QZNFRMXKQCIPQY-UHFFFAOYSA-N | $1.0\times10^{-3}$ | | Gharagheizi et al. (2010) | Q | 246 |
| benzenethiol | $2.9\times10^{-2}$ | | Duchowicz et al. (2020) | V | 186 |
| C$_6$H$_5$SH | $2.9\times10^{-2}$ | | HSDB (2015) | V | |
| (thiophenol) | $3.0\times10^{-2}$ | | Hine and Mookerjee (1975) | V | |
| [108-98-5] | $3.0\times10^{-2}$ | | Hine and Weimar (1965) | V | |
| RMVRSNDYEFQCLF-UHFFFAOYSA-N | $3.0\times10^{-2}$ | | Schüürmann (2000) | C | 21 |
| | $3.2\times10^{-1}$ | | Duchowicz et al. (2020) | Q | |
| | $4.1\times10^{-2}$ | | Hilal et al. (2008) | Q | |
| | $2.6\times10^{-2}$ | | Modarresi et al. (2007) | Q | 67 |
| | $2.9\times10^{-2}$ | | Yaffe et al. (2003) | Q | 248, 249 |
| | $1.6\times10^{-2}$ | | English and Carroll (2001) | Q | 230, 231 |
| | $1.8\times10^{-2}$ | | Katritzky et al. (1998) | Q | |
| | $1.0\times10^{-2}$ | | Nirmalakhandan et al. (1997) | Q | |
| | $3.0\times10^{-2}$ | | Suzuki et al. (1992) | Q | 232 |
| | $3.0\times10^{-2}$ | | Abraham et al. (1990) | ? | |
| methyl phenyl sulfide | $4.0\times10^{-2}$ | | Hine and Mookerjee (1975) | V | |
| C$_6$H$_5$SCH$_3$ | $4.1\times10^{-2}$ | | Hine and Weimar (1965) | V | |
| (thioanisole) | $5.8\times10^{-2}$ | | Hilal et al. (2008) | Q | |
| [100-68-5] | $4.3\times10^{-2}$ | | Yaffe et al. (2003) | Q | 248, 249 |
| HNKJADCVZUBCPG-UHFFFAOYSA-N | $2.3\times10^{-2}$ | | Nirmalakhandan et al. (1997) | Q | |
| | $4.0\times10^{-2}$ | | Suzuki et al. (1992) | Q | 232 |
| 2-methylbenzenethiol C$_7$H$_8$S (2-thiocresol) [137-06-4] LXUNZSDDXMPKLP-UHFFFAOYSA-N | $2.7\times10^{-2}$ | | HSDB (2015) | Q | 99 |
| 3-methylbenzenethiol C$_7$H$_8$S (3-thiocresol) [108-40-7] WRXOZRLZDJAYDR-UHFFFAOYSA-N | $2.7\times10^{-2}$ | | HSDB (2015) | Q | 99 |
| 4-methylbenzenethiol C$_7$H$_8$S (4-thiocresol) [106-45-6] WLHCBQAPPJAULW-UHFFFAOYSA-N | $2.7\times10^{-2}$ | | HSDB (2015) | Q | 99 |
| benzenemethanethiol C$_7$H$_8$S [100-53-8] UENWRTRMUIOCKN-UHFFFAOYSA-N | $4.7\times10^{-2}$ | | HSDB (2015) | Q | 99 |



Table A9.1: Sulfur (C, H, O, N, Cl, S) (... continued)

| Substance Formula (Trivial Name) [CAS Registry Number] InChIKey | $H_s^{cp}$ (at $T^{\ominus}$) $\left[\dfrac{\text{mol}}{\text{m}^3\,\text{Pa}}\right]$ | $\dfrac{\text{d}\ln H_s^{cp}}{\text{d}(1/T)}$ [K] | Reference | Type | Note |
|---|---|---|---|---|---|
| benzo[$b$]thiophene C$_8$H$_6$S [95-15-8] FCEHBMOGCRZNNI-UHFFFAOYSA-N | 4.1×10$^{-2}$ 3.6×10$^{-2}$ 3.0×10$^{-2}$ | | Mackay et al. (2006d) Mackay et al. (1995) Smith and Bomberger (1980) Yaws (1999) | V V V ? | 558  24 21, 12 |
| dibenzothiophene C$_{12}$H$_8$S [132-65-0] IYYZUPMFVPLQIF-UHFFFAOYSA-N | 2.9×10$^{-1}$ 2.9×10$^{-1}$ 2.3×10$^{-2}$ 2.3×10$^{-2}$ 2.3×10$^{-2}$ 1.2 3.3×10$^{-1}$ | | Duchowicz et al. (2020) HSDB (2015) Mackay et al. (2006d) Mackay et al. (1995) Smith and Bomberger (1980) Duchowicz et al. (2020) Parnis et al. (2015) | V V V V V Q Q | 186  24  369 |
| thianthrene C$_{12}$H$_8$S$_2$ [92-85-3] GVIJJXMXTUZIOD-UHFFFAOYSA-N | 4.0 | | Abraham et al. (2019) | Q | |
| 2-methyldibenzothiophene C$_{13}$H$_{10}$S [20928-02-3] VHUXLBLPAMBOJS-UHFFFAOYSA-N | 2.7×10$^{-1}$ | | Parnis et al. (2015) | Q | 369 |
| 2,3-dimethyldibenzothiophene C$_{14}$H$_{12}$S [31317-17-6] KEIKATUAMBEIQN-UHFFFAOYSA-N | 3.1×10$^{-1}$ | | Parnis et al. (2015) | Q | 369 |
| 2,8-dimethyldibenzothiophene C$_{14}$H$_{12}$S [1207-15-4] RRYWCJRYULRSJM-UHFFFAOYSA-N | 2.4×10$^{-1}$ | | Parnis et al. (2015) | Q | 369 |
| benzyl sulfide C$_{14}$H$_{14}$S [538-74-9] LUFPJJNWMYZRQE-UHFFFAOYSA-N | 1.9 | | HSDB (2015) | Q | 99 |
| 2,3,7-trimethyldibenzothiophene C$_{15}$H$_{14}$S HJSMFSZKADBZIO-UHFFFAOYSA-N | 2.7×10$^{-1}$ | | Parnis et al. (2015) | Q | 369 |
| 2,3,8-trimethyldibenzothiophene C$_{15}$H$_{14}$S QSOGKFSFLARZIE-UHFFFAOYSA-N | 2.7×10$^{-1}$ | | Parnis et al. (2015) | Q | 369 |
| 2,4,7-trimethyldibenzothiophene C$_{15}$H$_{14}$S [216983-03-8] YDWNRDFOOADVGC-UHFFFAOYSA-N | 1.9×10$^{-1}$ | | Parnis et al. (2015) | Q | 369 |





Table A9.1: Sulfur (C, H, O, N, Cl, S) (... continued)

| Substance<br>Formula<br>(Trivial Name)<br>[CAS Registry Number]<br>InChIKey | $H_s^{cp}$<br>(at $T^{\ominus}$)<br>$\left[\dfrac{\mathrm{mol}}{\mathrm{m^3\,Pa}}\right]$ | $\dfrac{\mathrm{d}\ln H_s^{cp}}{\mathrm{d}(1/T)}$<br><br>[K] | Reference | Type | Note |
|---|---|---|---|---|---|
| 4,6-diethyldibenzothiophene<br>$C_{16}H_{16}S$<br>[132034-91-4]<br>UMQGGSYHJPHWFV-UHFFFAOYSA-N | $1.3\times10^{-1}$ | | Parnis et al. (2015) | Q | 369 |
| 2-butyldibenzothiophene<br>$C_{16}H_{16}S$<br>[147792-31-2]<br>BEQMTJNPMZQPKZ-UHFFFAOYSA-N | $1.6\times10^{-1}$ | | Parnis et al. (2015) | Q | 369 |
| carbon oxide sulfide<br>OCS<br>(carbonyl sulfide)<br>[463-58-1]<br>JJWKPURADFRFRB-UHFFFAOYSA-N | $2.0\times10^{-4}$<br>$2.0\times10^{-4}$<br>$2.1\times10^{-4}$<br>$2.0\times10^{-4}$<br>$2.0\times10^{-4}$<br>$2.1\times10^{-4}$<br>$2.2\times10^{-4}$<br>$1.5\times10^{-4}$<br>$1.5\times10^{-4}$<br>$2.4\times10^{-4}$<br>$2.1\times10^{-4}$<br>$2.1\times10^{-4}$<br>$3.4\times10^{-4}$<br>$1.6\times10^{-5}$<br>$1.6\times10^{-5}$<br>$1.9\times10^{-4}$<br>$2.0\times10^{-4}$<br>5.4<br>$2.0\times10^{-4}$<br><br><br>$2.0\times10^{-4}$<br>$2.1\times10^{-4}$<br>$1.9\times10^{-4}$ | 3500<br>3500<br>3300<br>3500<br>3500<br>3000<br>2100<br>3800<br>3500<br><br>3300<br>3300<br><br><br><br><br><br><br><br>2900<br>3300<br><br>3000<br> | Burkholder et al. (2019)<br>Burkholder et al. (2015)<br>Warneck and Williams (2012)<br>Sander et al. (2011)<br>Sander et al. (2006)<br>Wilhelm et al. (1977)<br>De Bruyn et al. (1995b)<br>Johnson and Harrison (1986)<br>Hoyt (1982)<br>Stock and Kuß (1917)<br>Winkler (1907)<br>Winkler (1906)<br>Hempel (1901)<br>Duchowicz et al. (2020)<br>HSDB (2015)<br>Yaws (2003)<br>Hayer et al. (2022)<br>Duchowicz et al. (2020)<br>Gharagheizi et al. (2010)<br>Kühne et al. (2005)<br>Kühne et al. (2005)<br>Yaws (1999)<br>Yaws et al. (1999)<br>Yaws and Yang (1992) | L<br>L<br>L<br>L<br>L<br>L<br>M<br>M<br>M<br>M<br>M<br>M<br>M<br>V<br>V<br>X<br>Q<br>Q<br>Q<br>Q<br>?<br>?<br>?<br>? | <br><br><br><br><br><br><br>70<br>70<br><br><br><br>619<br>186<br><br>237<br>20<br><br>246<br><br><br>21<br>21<br>21 |
| methanesulfonic acid<br>$CH_3SO_3H$<br>(MSA)<br>[75-75-2]<br>AFVFQIVMOAPDHO-UHFFFAOYSA-N | <br>$7.3\times10^{3}$<br>$1.2\times10^{6}$<br>$5.0\times10^{5}$ | | Brimblecombe and Clegg (1988)<br>Wang et al. (2017)<br>Wang et al. (2017)<br>Wang et al. (2017) | T<br>Q<br>Q<br>Q | 813<br>80, 238<br>80, 239<br>80, 240 |
| MCM:CH3SO2OOH<br>$CH_4O_4S$<br>BENVNICSYZXTGE-UHFFFAOYSA-N | $1.9\times10^{5}$<br>$1.0\times10^{7}$<br>$1.8\times10^{4}$ | | Wang et al. (2017)<br>Wang et al. (2017)<br>Wang et al. (2017) | Q<br>Q<br>Q | 80, 238<br>80, 239<br>80, 240 |
| MCM:CH3SOOOH<br>$CH_4O_3S$<br>IZTYBZWDPIUSAG-UHFFFAOYSA-N | $4.3\times10^{7}$<br>$5.6\times10^{6}$<br>$2.7\times10^{3}$ | | Wang et al. (2017)<br>Wang et al. (2017)<br>Wang et al. (2017) | Q<br>Q<br>Q | 80, 238<br>80, 239<br>80, 240 |



Table A9.1: Sulfur (C, H, O, N, Cl, S) (... continued)

| Substance Formula (Trivial Name) [CAS Registry Number] InChIKey | $H_s^{cp}$ (at $T^{\ominus}$) $\left[\dfrac{\text{mol}}{\text{m}^3\,\text{Pa}}\right]$ | $\dfrac{\text{d}\ln H_s^{cp}}{\text{d}(1/T)}$ [K] | Reference | Type | Note |
|---|---|---|---|---|---|
| MCM:MSIA CH$_4$O$_2$S XNEFVTBPCXGIRX-UHFFFAOYSA-N | $1.6\times10^6$ $8.0\times10^3$ $3.6\times10^5$ | | Wang et al. (2017) Wang et al. (2017) Wang et al. (2017) | Q Q Q | 80, 238 80, 239 80, 240 |
| sulfuric acid, dimethyl ester C$_2$H$_6$O$_4$S (dimethyl sulfate) [77-78-1] VAYGXNSJCAHWJZ-UHFFFAOYSA-N | 6.9 2.5 | | Hilal et al. (2008) Yaws (1999) | Q ? | 21, 28 |
| dimethylsulfoxide CH$_3$SOCH$_3$ (DMSO) [67-68-5] IAZDPXIOMUYVGZ-UHFFFAOYSA-N | $9.8\times10^2$ $9.8\times10^2$ $8.0\times10^3$ $9.8\times10^2$ $9.8\times10^2$ $>9.9\times10^3$ $9.4\times10^2$ 4.4 4.4 $1.0\times10^4$ $1.4\times10^1$ $2.0\times10^2$ $1.3\times10^2$ $4.6\times10^2$ $3.9\times10^4$ $2.0\times10^5$ $4.3\times10^3$ $6.7\times10^3$ $6.5\times10^3$ | 7800 1300 8700 3100 4100 | Burkholder et al. (2019) Burkholder et al. (2015) Brockbank (2013) Sander et al. (2011) Sander et al. (2006) Lee and Zhou (1994) Watts and Brimblecombe (1987) Mackay et al. (2006d) Mackay et al. (1995) Bagno et al. (1991) Betterton (1992) Keshavarz et al. (2022) Duchowicz et al. (2020) Wang et al. (2017) Wang et al. (2017) Wang et al. (2017) Hilal et al. (2008) Kühne et al. (2005) Taft et al. (1985) Duchowicz et al. (2020) Kühne et al. (2005) Fogg and Sangster (2003) | L L L L L M M V V T C Q Q Q Q Q Q Q Q ? ? ? | 1 473 80, 238 80, 239 80, 240 185, 21 814 |
| dimethylsulfone CH$_3$SO$_2$CH$_3$ (DMSO2) [67-71-0] HHVIBTZHLRERCL-UHFFFAOYSA-N | $5.0\times10^{-3}$ $5.0\times10^{-3}$ 4.0 $3.3\times10^4$ $1.8\times10^5$ $>4.9\times10^2$ | | Mackay et al. (2006d) Mackay et al. (1995) Wang et al. (2017) Wang et al. (2017) Wang et al. (2017) De Bruyn et al. (1994) | V V Q Q Q E | 80, 238 80, 239 80, 240 |
| mercaptoacetic acid C$_2$H$_4$O$_2$S [68-11-1] CWERGRDVMFNCDR-UHFFFAOYSA-N | $5.2\times10^2$ | | HSDB (2015) | Q | 99 |
| 2-mercaptoethanol C$_2$H$_6$OS [60-24-2] DGVVWUTYPXICAM-UHFFFAOYSA-N | $5.5\times10^1$ $5.5\times10^1$ $1.0\times10^2$ | | Duchowicz et al. (2020) HSDB (2015) Duchowicz et al. (2020) | V V Q | 186 |



Table A9.1: Sulfur (C, H, O, N, Cl, S) (. . . continued)

| Substance<br>Formula<br>(Trivial Name)<br>[CAS Registry Number]<br>InChIKey | $H_s^{cp}$<br>(at $T^\ominus$)<br>$\left[\dfrac{\mathrm{mol}}{\mathrm{m^3\,Pa}}\right]$ | $\dfrac{\mathrm{d}\ln H_s^{cp}}{\mathrm{d}(1/T)}$<br><br>[K] | Reference | Type | Note |
|---|---|---|---|---|---|
| methanesulfonic acid, methyl ester<br>$C_2H_6O_3S$<br>[66-27-3]<br>MBABOKRGFJTBAE-UHFFFAOYSA-N | 2.5 | | HSDB (2015) | Q | 99 |
| MCM:CH3SCH2OOH<br>$C_2H_6O_2S$<br>LDZPECMXMDESRV-UHFFFAOYSA-N | $1.1\times10^2$<br>$1.4\times10^2$<br>9.8 | | Wang et al. (2017)<br>Wang et al. (2017)<br>Wang et al. (2017) | Q<br>Q<br>Q | 80, 238<br>80, 239<br>80, 240 |
| MCM:DMSO2OOH<br>$C_2H_6O_4S$<br>GCLQZLAJAMWHPS-UHFFFAOYSA-N | $2.6\times10^5$<br>$6.5\times10^6$<br>$6.8\times10^5$ | | Wang et al. (2017)<br>Wang et al. (2017)<br>Wang et al. (2017) | Q<br>Q<br>Q | 80, 238<br>80, 239<br>80, 240 |
| MCM:CH3SCH2OH<br>$C_2H_6OS$<br>ZSSFPSNLAUYOFG-UHFFFAOYSA-N | 4.2<br>$7.6\times10^1$<br>$1.1\times10^1$ | | Wang et al. (2017)<br>Wang et al. (2017)<br>Wang et al. (2017) | Q<br>Q<br>Q | 80, 238<br>80, 239<br>80, 240 |
| MCM:DMSO2OH<br>$C_2H_6O_3S$<br>ICHBUPLXTAHKLA-UHFFFAOYSA-N | $1.0\times10^4$<br>$5.6\times10^6$<br>$5.8\times10^6$ | | Wang et al. (2017)<br>Wang et al. (2017)<br>Wang et al. (2017) | Q<br>Q<br>Q | 80, 238<br>80, 239<br>80, 240 |
| MCM:CH3SCHO<br>$C_2H_4OS$<br>LFJRGYNYRORDDM-UHFFFAOYSA-N | 1.3<br>$1.5\times10^{-1}$<br>$4.8\times10^{-2}$ | | Wang et al. (2017)<br>Wang et al. (2017)<br>Wang et al. (2017) | Q<br>Q<br>Q | 80, 238<br>80, 239<br>80, 240 |
| MCM:CH3SO2CHO<br>$C_2H_4O_3S$<br>LUGQISNSRPQOGS-UHFFFAOYSA-N | $3.1\times10^3$<br>$3.6\times10^2$<br>$2.2\times10^2$ | | Wang et al. (2017)<br>Wang et al. (2017)<br>Wang et al. (2017) | Q<br>Q<br>Q | 80, 238<br>80, 239<br>80, 240 |
| mercaptoacetic acid, methyl ester<br>$C_3H_6O_2S$<br>(methyl thioglycolate)<br>[2365-48-2]<br>MKIJJIMOAABWGF-UHFFFAOYSA-N | 1.6 | | HSDB (2015) | Q | 99 |
| 1,3-propane sultone<br>$C_3H_6O_3S$<br>[1120-71-4]<br>FSSPGSAQUIYDCN-UHFFFAOYSA-N | $1.6\times10^3$ | | Ebert et al. (2023) | ? | 316 |
| methanesulfonic acid, ethyl ester<br>$C_3H_8O_3S$<br>[62-50-0]<br>PLUBXMRUUVWRLT-UHFFFAOYSA-N | $3.8\times10^1$<br>$2.8\times10^{-2}$<br>1.8 | | Duchowicz et al. (2020)<br>Duchowicz et al. (2020)<br>HSDB (2015) | V<br>Q<br>Q | 186<br><br>99 |
| 2,3-dimercapto-1-propanol<br>$C_3H_8OS_2$<br>[59-52-9]<br>WQABCVAJNWAXTE-UHFFFAOYSA-N | $1.1\times10^3$<br>$4.1\times10^3$ | | Duchowicz et al. (2020)<br>Duchowicz et al. (2020) | V<br>Q | 186 |



Table A9.1: Sulfur (C, H, O, N, Cl, S) (... continued)

| Substance Formula (Trivial Name) [CAS Registry Number] InChIKey | $H_s^{cp}$ (at $T^{\ominus}$) $\left[\dfrac{\mathrm{mol}}{\mathrm{m^3\,Pa}}\right]$ | $\dfrac{\mathrm{d}\ln H_s^{cp}}{\mathrm{d}(1/T)}$ [K] | Reference | Type | Note |
|---|---|---|---|---|---|
| divinyl sulfoxide $C_4H_6OS$ (vinyl sulfoxide) [1115-15-7] HQSMEHLVLOGBCK-UHFFFAOYSA-N | $2.5 \times 10^1$ | | HSDB (2015) | Q | 99 |
| divinyl sulphone $C_4H_6O_2S$ [77-77-0] AFOSIXZFDONLBT-UHFFFAOYSA-N | $2.0 \times 10^{-1}$ | | HSDB (2015) | Q | 99 |
| 2,5-dihydrothiophene sulfone $C_4H_6O_2S$ (2,5-dihydrothiophene 1,1-dioxide) [77-79-2] MBDNRNMVTZADMQ-UHFFFAOYSA-N | 2.3 | | HSDB (2015) | Q | 99 |
| thiodiacetic acid $C_4H_6O_4S$ [123-93-3] UVZICZIVKIMRNE-UHFFFAOYSA-N | $2.2 \times 10^8$ | | HSDB (2015) | Q | 99 |
| thiophene, tetrahydro-, 1,1-dioxide $C_4H_8O_2S$ (sulfolane) [126-33-0] HXJUTPCZVOIRIF-UHFFFAOYSA-N | 2.1 | | HSDB (2015) | Q | 99 |
| 2-(ethylthio)ethanol $C_4H_{10}OS$ [110-77-0] LNRIEBFNWGMXKP-UHFFFAOYSA-N | $1.9 \times 10^2$ | | HSDB (2015) | Q | 99 |
| thiodiglycol $C_4H_{10}O_2S$ [111-48-8] YODZTKMDCQEPHD-UHFFFAOYSA-N | $5.2 \times 10^3$ | | HSDB (2015) | Q | 99 |
| methanesulfonic acid, 1-methylethyl ester $C_4H_{10}O_3S$ [926-06-7] SWWHCQCMVCPLEQ-UHFFFAOYSA-N | 1.4 | | HSDB (2015) | Q | 99 |
| diethyl sulfate $C_4H_{10}O_4S$ [64-67-5] DENRZWYUOJLTMF-UHFFFAOYSA-N | 1.6 | | Ebert et al. (2023) | ? | 316 |



Table A9.1: Sulfur (C, H, O, N, Cl, S) (... continued)

| Substance<br>Formula<br>(Trivial Name)<br>[CAS Registry Number]<br>InChIKey | $H_s^{cp}$<br>(at $T^{\ominus}$)<br>$\left[\dfrac{\text{mol}}{\text{m}^3\,\text{Pa}}\right]$ | $\dfrac{\text{d}\ln H_s^{cp}}{\text{d}(1/T)}$<br><br>[K] | Reference | Type | Note |
|---|---|---|---|---|---|
| S-methyl butanethioate<br>$C_5H_{10}OS$<br>[2432-51-1]<br>GRLJIIJNZJVMGP-UHFFFAOYSA-N | $2.6\times10^{-2}$ | | Souchon et al. (2004) | M | |
| 4-hydroxybenzenesulfonic acid<br>$C_6H_6O_4S$<br>[98-67-9]<br>FEPBITJSIHRMRT-UHFFFAOYSA-N | $3.8\times10^{7}$ | | HSDB (2015) | Q | 99 |
| benzenesulfonic acid<br>$C_6H_6O_3S$<br>[98-11-3]<br>SRSXLGNVWSONIS-UHFFFAOYSA-N | $3.9\times10^{3}$ | | HSDB (2015) | Q | 99 |
| dimethipin<br>$C_6H_{10}O_4S_2$<br>[55290-64-7]<br>PHVNLLCAQHGNKU-UHFFFAOYSA-N | $4.3\times10^{5}$<br>$7.0\times10^{5}$<br>$4.3\times10^{5}$ | | Duchowicz et al. (2020)<br>Duchowicz et al. (2020)<br>MacBean (2012a) | V<br>Q<br>? | 186 |
| 4-methylbenzenesulfonic acid<br>$C_7H_8O_3S$<br>[104-15-4]<br>JOXIMZWYDAKGHI-UHFFFAOYSA-N | $3.6\times10^{3}$ | | HSDB (2015) | Q | 99 |
| phenylmethanesulfonic acid<br>$C_7H_8O_3S$<br>(benzylsulfonic acid)<br>[100-87-8]<br>NIXKBAZVOQAHGC-UHFFFAOYSA-N | $9.9\times10^{3}$ | | HSDB (2015) | Q | 99 |
| isoprothiolane<br>$C_{12}H_{18}O_4S_2$<br>[50512-35-1]<br>UFHLMYOGRXOCSL-UHFFFAOYSA-N | $9.8\times10^{-2}$<br>$2.8\times10^{3}$ | | Duchowicz et al. (2020)<br>Duchowicz et al. (2020) | V<br>Q | 186 |
| 4,4'-sulfonyldiphenol<br>$C_{12}H_{10}O_4S$<br>(bisphenol S)<br>[80-09-1]<br>VPWNQTHUCYMVMZ-UHFFFAOYSA-N | $3.7\times10^{9}$ | | HSDB (2015) | Q | 447 |
| benfuresate<br>$C_{12}H_{16}O_4S$<br>[68505-69-1]<br>QGQSRQPXXMTJCM-UHFFFAOYSA-N | $3.7\times10^{2}$ | | Ebert et al. (2023) | ? | 318 |



Table A9.1: Sulfur (C, H, O, N, Cl, S) (. . . continued)

| Substance Formula (Trivial Name) [CAS Registry Number] InChIKey | $H_s^{cp}$ (at $T^\ominus$) $\left[\dfrac{\text{mol}}{\text{m}^3\,\text{Pa}}\right]$ | $\dfrac{\text{d}\ln H_s^{cp}}{\text{d}(1/T)}$ [K] | Reference | Type | Note |
|---|---|---|---|---|---|
| lauryl sulfate $C_{12}H_{26}O_4S$ (dodecyl sulfate) [151-41-7] MOTZDAYCYVMXPC-UHFFFAOYSA-N | $5.5\times10^1$ | | HSDB (2015) | Q | 99 |
| ethofumesate $C_{13}H_{18}O_5S$ [26225-79-6] IRCMYGHHKLLGHV-UHFFFAOYSA-N | $2.7\times10^2$ $2.7\times10^2$ $4.5\times10^1$ 7.7 $1.1\times10^2$ $2.7\times10^2$ | | Duchowicz et al. (2020) HSDB (2015) Barcelo and Hennion (1997) Duchowicz et al. (2020) Goodarzi et al. (2010) Maniere et al. (2011) | V V X Q Q ? | 186 567 568, 569 165 |
| 1,1'-sulfonylbis(4-(1-methylethyl)-benzene $C_{18}H_{22}O_2S$ [57913-35-6] PKQWGBLKCJIMDP-UHFFFAOYSA-N | $1.0\times10^1$ $2.9\times10^3$ $6.7\times10^4$ $3.1\times10^2$ | | Zhang et al. (2010) Zhang et al. (2010) Zhang et al. (2010) Zhang et al. (2010) | Q Q Q Q | 287, 288 287, 289 287, 290 287, 291 |
| propargite $C_{19}H_{26}O_4S$ [2312-35-8] ZYHMJXZULPZUED-UHFFFAOYSA-N | $1.5\times10^1$ $1.5\times10^1$ $1.0\times10^2$ | | Duchowicz et al. (2020) HSDB (2015) Duchowicz et al. (2020) | V V Q | 186 |
| kadethrin $C_{23}H_{24}O_4S$ [58769-20-3] UGWALRUNBSBTGI-QJLCOAGJSA-N | $1.2\times10^4$ | | HSDB (2015) | Q | 99 |
| spironolactone $C_{24}H_{32}O_4S$ [52-01-7] LXMSZDCAJNLERA-PJKOONHHSA-N | $9.0\times10^4$ | | HSDB (2015) | Q | 99 |
| 2,2'-thiobis(4-(1,1,3,3-tetramethylbutyl)phenol) $C_{28}H_{42}O_2S$ [3294-03-9] WQYFETFRIRDUPJ-UHFFFAOYSA-N | $4.5\times10^5$ $1.2\times10^4$ $2.2\times10^7$ $1.8\times10^5$ | | Zhang et al. (2010) Zhang et al. (2010) Zhang et al. (2010) Zhang et al. (2010) | Q Q Q Q | 287, 288 287, 289 287, 290 287, 291 |
| propanoic acid, 3,3'-thiobis-, didodecyl ester $C_{30}H_{58}O_4S$ (dilauryl thiodipropionate) [123-28-4] GHKOFFNLGXMVNJ-UHFFFAOYSA-N | 2.5 | | HSDB (2015) | Q | 447 |
| dioctadecyl 3,3'-thiodipropionate $C_{42}H_{82}O_4S$ [693-36-7] PWWSSIYVTQUJQQ-UHFFFAOYSA-N | $8.2\times10^{-2}$ | | HSDB (2015) | Q | 447 |





Table A9.1: Sulfur (C, H, O, N, Cl, S) (...continued)

| Substance<br>Formula<br>(Trivial Name)<br>[CAS Registry Number]<br><small>InChIKey</small> | $H_s^{cp}$<br>(at $T^{\ominus}$)<br>$\left[\dfrac{\text{mol}}{\text{m}^3\,\text{Pa}}\right]$ | $\dfrac{\text{d}\ln H_s^{cp}}{\text{d}(1/T)}$<br><br>[K] | Reference | Type | Note |
|---|---|---|---|---|---|
| methyl isothiocyanate<br>$CH_3NCS$<br>[556-61-6]<br><small>LGDSHSYDSCRFAB-UHFFFAOYSA-N</small> | $1.7\times10^{-1}$<br>$1.7\times10^{-1}$<br>$1.7\times10^{-1}$<br>$1.6\times10^{-1}$<br>$2.2\times10^{-1}$<br>$2.2\times10^{-1}$<br>$2.1$<br>$1.8\times10^{-2}$ | | Burkholder et al. (2019)<br>Burkholder et al. (2015)<br>Sander et al. (2011)<br>Worthington and Wade (2007)<br>Duchowicz et al. (2020)<br>HSDB (2015)<br>Duchowicz et al. (2020)<br>Modarresi et al. (2007) | L<br>L<br>L<br>M<br>V<br>V<br>Q<br>Q | <br><br><br><br>186<br><br><br>67 |
| thiourea<br>$CH_4N_2S$<br>[62-56-6]<br><small>UMGDCJDMYOKAJW-UHFFFAOYSA-N</small> | $5.0\times10^{3}$<br>$4.9\times10^{3}$<br>$1.7\times10^{5}$ | | Duchowicz et al. (2020)<br>HSDB (2015)<br>Duchowicz et al. (2020) | V<br>V<br>Q | 186 |
| hydrazinecarbothioamide<br>$CH_5N_3S$<br>(1-amino-2-thiourea)<br>[79-19-6]<br><small>BRWIZMBXBAOCCF-UHFFFAOYSA-N</small> | $1.5\times10^{4}$ | | HSDB (2015) | Q | 99 |
| thiocyanic acid, methyl ester<br>$C_2H_3NS$<br>[556-64-9]<br><small>VYHVQEYOFIYNJP-UHFFFAOYSA-N</small> | $2.2\times10^{-1}$ | | HSDB (2015) | Q | 99 |
| ethanethioamide<br>$C_2H_5NS$<br>(thioacetamide)<br>[62-55-5]<br><small>YUKQRDCYNOVPGJ-UHFFFAOYSA-N</small> | $1.5$ | | HSDB (2015) | Q | 99 |
| mercaptamine<br>$C_2H_7NS$<br>(cysteamine)<br>[60-23-1]<br><small>UFULAYFCSOUIOV-UHFFFAOYSA-N</small> | $2.7\times10^{1}$ | | HSDB (2015) | Q | 99 |
| thiocyanic acid, ethyl ester<br>$C_3H_5NS$<br>[542-90-5]<br><small>WFCLYEAZTHWNEH-UHFFFAOYSA-N</small> | $1.7\times10^{-1}$ | | HSDB (2015) | Q | 99 |
| 2-imidazolidinethione<br>$C_3H_6N_2S$<br>(ethylene thiourea)<br>[96-45-7]<br><small>PDQAZBWRQCGBEV-UHFFFAOYSA-N</small> | $2.9\times10^{1}$ | | HSDB (2015) | Q | 99 |



Table A9.1: Sulfur (C, H, O, N, Cl, S) (…continued)

| Substance Formula (Trivial Name) [CAS Registry Number] InChIKey | $H_s^{cp}$ (at $T^\ominus$) $\left[\dfrac{\mathrm{mol}}{\mathrm{m^3\,Pa}}\right]$ | $\dfrac{\mathrm{d}\ln H_s^{cp}}{\mathrm{d}(1/T)}$ [K] | Reference | Type | Note |
|---|---|---|---|---|---|
| ethylthiourea $C_3H_8N_2S$ [625-53-6] GMEHFXXZSWDEDB-UHFFFAOYSA-N | $4.2\times10^2$ | | HSDB (2015) | Q | 99 |
| allyl isothiocyanate $C_4H_5NS$ [57-06-7] ZOJBYZNEUISWFT-UHFFFAOYSA-N | $2.1\times10^{-2}$ $4.1\times10^{-2}$ $4.1\times10^{-2}$ 2.7 $2.3\times10^{-1}$ | | Souchon et al. (2004) Duchowicz et al. (2020) HSDB (2015) Duchowicz et al. (2020) Modarresi et al. (2007) | M V V Q Q | 186 67 |
| dazomet $C_5H_{10}N_2S_2$ [533-74-4] QAYICIQNSGETAS-UHFFFAOYSA-N | $2.0\times10^4$ $4.6\times10^4$ $3.7\times10^4$ $3.7\times10^5$ $2.0\times10^4$ | | Duchowicz et al. (2020) Mackay et al. (2006d) MacBean (2012b) Duchowicz et al. (2020) Maniere et al. (2011) | V V X Q ? | 186 350 12, 165 |
| N,N'-diethylthiourea $C_5H_{12}N_2S$ [105-55-5] FLVIGYVXZHLUHP-UHFFFAOYSA-N | $1.4\times10^2$ | | HSDB (2015) | Q | 99 |
| tetramethylthiourea $C_5H_{12}N_2S$ [2782-91-4] MNOILHPDHOHILI-UHFFFAOYSA-N | $8.5\times10^2$ | | HSDB (2015) | Q | 99 |
| thiram $C_6H_{12}N_2S_4$ [137-26-8] KUAZQDVKQLNFPE-UHFFFAOYSA-N | $9.3\times10^1$ $3.0\times10^1$ | | Mackay et al. (2006d) MacBean (2012b) | V X | 350 |
| bis(dimethylthiocarbamyl) sulfide $C_6H_{12}N_2S_3$ (bis(dimethylthiocarbamoyl) sulfide) [97-74-5] REQPQFUJGGOFQL-UHFFFAOYSA-N | $5.8\times10^{-1}$ | | HSDB (2015) | Q | 99 |
| benzothiazole $C_7H_5NS$ [95-16-9] IOJUPLGTWVMSFF-UHFFFAOYSA-N | $2.7\times10^1$ $2.7\times10^1$ 2.8 $1.1\times10^1$ 2.0 | | HSDB (2015) Zhang et al. (2010) Zhang et al. (2010) Zhang et al. (2010) Zhang et al. (2010) | Q Q Q Q Q | 99 287, 288 287, 289 287, 290 287, 291 |
| 2-mercaptobenzothiazole $C_7H_5NS_2$ [149-30-4] YXIWHUQXZSMYRE-UHFFFAOYSA-N | $2.7\times10^2$ $2.7\times10^2$ $2.8\times10^3$ $2.2\times10^2$ $2.1\times10^1$ | | HSDB (2015) Zhang et al. (2010) Zhang et al. (2010) Zhang et al. (2010) Zhang et al. (2010) | Q Q Q Q Q | 99 287, 288 287, 289 287, 290 287, 291 |





Table A9.1: Sulfur (C, H, O, N, Cl, S) (...continued)

| Substance<br>Formula<br>(Trivial Name)<br>[CAS Registry Number]<br>InChIKey | $H_s^{cp}$<br>(at $T^{\ominus}$)<br>$\left[\dfrac{\text{mol}}{\text{m}^3\,\text{Pa}}\right]$ | $\dfrac{\text{d}\ln H_s^{cp}}{\text{d}(1/T)}$<br><br>[K] | Reference | Type | Note |
|---|---|---|---|---|---|
| 2-benzothiazolamine<br>C$_7$H$_6$N$_2$S<br>[136-95-8]<br>UHGULLIUJBCTEF-UHFFFAOYSA-N | $7.6\times10^4$ | | HSDB (2015) | Q | 99 |
| phenylthiourea<br>C$_7$H$_8$N$_2$S<br>[103-85-5]<br>FULZLIGZKMKICU-UHFFFAOYSA-N | $9.9\times10^4$ | | HSDB (2015) | Q | 99 |
| aziprotryn<br>C$_7$H$_{11}$N$_7$S<br>[4658-28-0]<br>AFIIBUOYKYSPKB-UHFFFAOYSA-N | $4.0\times10^2$<br>$9.2\times10^2$ | | Abraham et al. (2007)<br>MacBean (2012a) | Q<br>? | |
| isothiocyanatobenzene<br>C$_7$H$_5$NS<br>[103-72-0]<br>QKFJKGMPGYROCL-UHFFFAOYSA-N | $3.3\times10^{-3}$<br>5.1 | | Duchowicz et al. (2020)<br>Duchowicz et al. (2020) | V<br>Q | 186 |
| simetryn<br>C$_8$H$_{15}$N$_5$S<br>[1014-70-6]<br>MGLWZSOBALDPEK-UHFFFAOYSA-N | $2.2\times10^4$<br>$2.2\times10^4$<br>$3.8\times10^6$<br>$2.9\times10^4$<br>$1.0\times10^4$ | | Duchowicz et al. (2020)<br>Mackay et al. (2006d)<br>Duchowicz et al. (2020)<br>Hilal et al. (2008)<br>Abraham et al. (2007) | V<br>V<br>Q<br>Q<br>Q | 186 |
| desmetryn<br>C$_8$H$_{15}$N$_5$S<br>[1014-69-3]<br>HCRWJJJUKUVORR-UHFFFAOYSA-N | $2.1\times10^4$<br>$5.0\times10^7$<br>$2.2\times10^4$<br>$2.0\times10^4$<br>$1.4\times10^9$<br>$3.9\times10^7$<br>$2.1\times10^4$ | | HSDB (2015)<br>Delgado and Alderete (2003)<br>Hilal et al. (2008)<br>Abraham et al. (2007)<br>Delgado and Alderete (2003)<br>Delgado and Alderete (2003)<br>MacBean (2012a) | V<br>C<br>Q<br>Q<br>Q<br>Q<br>? | |
| thioquinox<br>C$_9$H$_4$N$_2$S$_3$<br>[93-75-4]<br>ILERPRJWJPJZDN-UHFFFAOYSA-N | $1.3\times10^2$ | | HSDB (2015) | Q | 99 |
| thiocyanic acid,<br>(2-benzothiazolylthio)methyl ester<br>C$_9$H$_6$N$_2$S$_3$<br>[21564-17-0]<br>TUBQDCKAWGHZPF-UHFFFAOYSA-N | $1.5\times10^6$ | | HSDB (2015) | Q | 99 |



Table A9.1: Sulfur (C, H, O, N, Cl, S) (. . . continued)

| Substance Formula (Trivial Name) [CAS Registry Number] InChIKey | $H_s^{cp}$ (at $T^{\ominus}$) $\left[ \dfrac{\text{mol}}{\text{m}^3\,\text{Pa}} \right]$ | $\dfrac{\text{d} \ln H_s^{cp}}{\text{d}(1/T)}$ [K] | Reference | Type | Note |
|---|---|---|---|---|---|
| ametryn $C_9H_{17}N_5S$ [834-12-8] RQVYBGPQFYCBGX-UHFFFAOYSA-N | $2.1\times10^1$ $4.1\times10^3$ $8.1\times10^3$ $8.3\times10^3$ $8.2\times10^1$ $4.1\times10^3$ $1.3\times10^2$ $1.2\times10^4$ $5.1\times10^3$ $8.9\times10^7$ $1.1\times10^7$ | | Chao et al. (2017) HSDB (2015) Mackay et al. (2006d) Suntio et al. (1988) Barcelo and Hennion (1997) Delgado and Alderete (2003) Goodarzi et al. (2010) Hilal et al. (2008) Abraham et al. (2007) Delgado and Alderete (2003) Delgado and Alderete (2003) | M V V V X C Q Q Q Q Q | 12 567 568 |
| cimetidine $C_{10}H_{16}N_6S$ [51481-61-9] AQIXAKUUQRKLND-UHFFFAOYSA-N | $1.0\times10^{10}$ | | HSDB (2015) | Q | 99 |
| prometryn $C_{10}H_{19}N_5S$ [7287-19-6] AAEVYOVXGOFMJO-UHFFFAOYSA-N | $8.2\times10^2$ $2.0\times10^3$ $2.0\times10^3$ $2.0\times10^1$ $2.9\times10^6$ $7.6\times10^2$ $2.5\times10^2$ $7.5\times10^2$ $2.5\times10^3$ $5.1\times10^6$ $1.4\times10^6$ | | HSDB (2015) Mackay et al. (2006d) Suntio et al. (1988) Barcelo and Hennion (1997) Delgado and Alderete (2003) Delgado and Alderete (2003) Goodarzi et al. (2010) Hilal et al. (2008) Abraham et al. (2007) Delgado and Alderete (2003) Delgado and Alderete (2003) | V V V X C C Q Q Q Q Q | 12 567 568 |
| terbutryn $C_{10}H_{19}N_5S$ [886-50-0] IROINLKCQGIITA-UHFFFAOYSA-N | $4.7\times10^2$ $7.0\times10^2$ $7.7\times10^2$ $7.6$ $1.2\times10^6$ $8.7\times10^2$ $1.7\times10^2$ $4.5\times10^3$ $1.6\times10^3$ $5.1\times10^6$ $1.4\times10^6$ | | HSDB (2015) Mackay et al. (2006d) Suntio et al. (1988) Barcelo and Hennion (1997) Delgado and Alderete (2003) Delgado and Alderete (2003) Goodarzi et al. (2010) Hilal et al. (2008) Abraham et al. (2007) Delgado and Alderete (2003) Delgado and Alderete (2003) | V V V X C C Q Q Q Q Q | 12 567 568 |
| disulfiram $C_{10}H_{20}N_2S_4$ [97-77-8] AUZONCFQVSMFAP-UHFFFAOYSA-N | $1.2\times10^{-1}$ | | HSDB (2015) | Q | 99 |



Table A9.1: Sulfur (C, H, O, N, Cl, S) (…continued)

| Substance Formula (Trivial Name) [CAS Registry Number] InChIKey | $H_s^{cp}$ (at $T^{\ominus}$) $\left[\dfrac{\mathrm{mol}}{\mathrm{m^3\,Pa}}\right]$ | $\dfrac{\mathrm{d\ln}H_s^{cp}}{\mathrm{d}(1/T)}$ [K] | Reference | Type | Note |
|---|---|---|---|---|---|
| 1-naphthalenylthiourea $C_{11}H_{10}N_2S$ [86-88-4] PIVQQUNOTICCSA-UHFFFAOYSA-N | $1.2\times10^3$ | | HSDB (2015) | Q | 99 |
| 4,4'-thiobisbenzenamine $C_{12}H_{12}N_2S$ (bis(4-aminophenyl) sulfide) [139-65-1] ICNFHJVPAJKPHW-UHFFFAOYSA-N | $2.5\times10^6$ | | HSDB (2015) | Q | 99 |
| dipropetryn $C_{11}H_{19}N_5S$ [4147-51-7] NPWMZOGDXOFZIN-UHFFFAOYSA-N | $6.0\times10^2$ $1.6\times10^3$ $6.5\times10^2$ | | Hilal et al. (2008) Abraham et al. (2007) MacBean (2012a) | Q Q ? | |
| dimethametryn $C_{11}H_{21}N_5S$ [22936-75-0] IKYICRRUVNIHPP-UHFFFAOYSA-N | $8.2\times10^3$ $1.0\times10^3$ | | Hilal et al. (2008) Abraham et al. (2007) | Q Q | |
| phenothiazine $C_{12}H_9NS$ [92-84-2] WJFKNYWRSNBZNX-UHFFFAOYSA-N | $1.0\times10^2$ $3.5\times10^2$ $3.5\times10^2$ $6.9\times10^2$ $9.7\times10^1$ $4.3\times10^2$ | | Abraham et al. (2019) HSDB (2015) Zhang et al. (2010) Zhang et al. (2010) Zhang et al. (2010) Zhang et al. (2010) | Q Q Q Q Q Q | 99 287, 288 287, 289 287, 290 287, 291 |
| cymiazole $C_{12}H_{14}N_2S$ [61676-87-7] YUAUPYJCVKNAEC-UHFFFAOYSA-N | $1.5\times10^2$ | | Ebert et al. (2023) | ? | 318 |
| 2,2'-dithiobisbenzothiazole $C_{14}H_8N_2S_4$ (2,2'-dibenzothiazyl disulfide) [120-78-5] AFZSMODLJJCVPP-UHFFFAOYSA-N | $4.2\times10^7$ | | HSDB (2015) | Q | 99 |
| methapyrilene $C_{14}H_{19}N_3S$ [91-80-5] HNJJXZKZRAWDPF-UHFFFAOYSA-N | $3.6\times10^1$ | | HSDB (2015) | V | |
| olanzapine $C_{17}H_{20}N_4S$ [132539-06-1] KVWDHTXUZHCGIO-UHFFFAOYSA-N | $1.3\times10^9$ | | HSDB (2015) | Q | 99 |



Table A9.1: Sulfur (C, H, O, N, Cl, S) (. . . continued)

| Substance Formula (Trivial Name) [CAS Registry Number] InChIKey | $H_s^{cp}$ (at $T^{\ominus}$) $\left[\dfrac{\text{mol}}{\text{m}^3\,\text{Pa}}\right]$ | $\dfrac{\text{d}\ln H_s^{cp}}{\text{d}(1/T)}$ [K] | Reference | Type | Note |
|---|---|---|---|---|---|
| N-(1,1-dimethylethyl)bis(2-benzothiazolesulfen)amide | $2.4\times10^8$ | | Zhang et al. (2010) | Q | 287, 288 |
| $C_{18}H_{17}N_3S_4$ | $1.7\times10^3$ | | Zhang et al. (2010) | Q | 287, 289 |
| [3741-80-8] | $2.3\times10^3$ | | Zhang et al. (2010) | Q | 287, 290 |
| VILGDADBAQFRJE-UHFFFAOYSA-N | $3.9\times10^8$ | | Zhang et al. (2010) | Q | 287, 291 |
| buthiobate $C_{21}H_{28}N_2S_2$ [51308-54-4] ZZVVDIVWGXTDRQ-UHFFFAOYSA-N | $3.4\times10^1$ | | Ebert et al. (2023) | ? | 318 |
| MCM:CH3SO4NO2 | $7.8\times10^3$ | | Wang et al. (2017) | Q | 80, 238 |
| $CH_3NO_6S$ | $1.6\times10^4$ | | Wang et al. (2017) | Q | 80, 239 |
| IMLJWBRSRKRPAQ-UHFFFAOYSA-N | $3.2\times10^{-2}$ | | Wang et al. (2017) | Q | 80, 240 |
| MCM:CH3SOO2NO2 | $1.7\times10^6$ | | Wang et al. (2017) | Q | 80, 238 |
| $CH_3NO_5S$ | $3.9\times10^8$ | | Wang et al. (2017) | Q | 80, 239 |
| PGTDFOBGUOOGGZ-UHFFFAOYSA-N | $8.3\times10^{-2}$ | | Wang et al. (2017) | Q | 80, 240 |
| taurine $C_2H_7NO_3S$ [107-35-7] XOAAWQZATWQOTB-UHFFFAOYSA-N | $5.8\times10^6$ | | HSDB (2015) | Q | 447 |
| 2-amino-5-nitrothiazole $C_3H_3N_3O_2S$ [121-66-4] MIHADVKEHAFNPG-UHFFFAOYSA-N | $1.9\times10^6$ | | HSDB (2015) | Q | 99 |
| N-(aminothioxomethyl)acetamide $C_3H_6N_2OS$ (1-acetyl-2-thiourea) [591-08-2] IPCRBOOJBPETMF-UHFFFAOYSA-N | $3.8\times10^5$ | | HSDB (2015) | Q | 99 |
| acesulfame $C_4H_5NO_4S$ [33665-90-6] YGCFIWIQZPHFLU-UHFFFAOYSA-N | $1.0\times10^3$ | | HSDB (2015) | Q | 99 |
| methomyl | $2.0\times10^2$ | | Chao et al. (2017) | M | |
| $C_5H_{10}N_2O_2S$ | $5.0\times10^5$ | | Duchowicz et al. (2020) | V | 186 |
| [16752-77-5] | $5.2\times10^5$ | | HSDB (2015) | V | |
| UHXUZOCRWCRNSJ-UHFFFAOYSA-N | $5.3\times10^4$ | | Mackay et al. (2006d) | V | |
| | $1.5\times10^4$ | | Suntio et al. (1988) | V | 12 |
| | $1.5\times10^2$ | | Barcelo and Hennion (1997) | X | 567 |
| | $1.0\times10^4$ | | Duchowicz et al. (2020) | Q | |
| | $1.3\times10^2$ | | Goodarzi et al. (2010) | Q | 568, 569 |



Table A9.1: Sulfur (C, H, O, N, Cl, S) (...continued)

| Substance<br>Formula<br>(Trivial Name)<br>[CAS Registry Number]<br>InChIKey | $H_s^{cp}$<br>(at $T^\ominus$)<br>$\left[\dfrac{\mathrm{mol}}{\mathrm{m}^3\,\mathrm{Pa}}\right]$ | $\dfrac{\mathrm{d}\ln H_s^{cp}}{\mathrm{d}(1/T)}$<br><br>[K] | Reference | Type | Note |
|---|---|---|---|---|---|
| 4-aminobenzenesulfonic acid<br>$C_6H_7NO_3S$<br>(sulfanilic acid)<br>[121-57-3]<br>HVBSAKJJOYLTQU-UHFFFAOYSA-N | $1.1\times10^7$ | | HSDB (2015) | Q | 447 |
| sulfanilamide<br>$C_6H_8N_2O_2S$<br>[63-74-1]<br>FDDDEECHVMSUSB-UHFFFAOYSA-N | $6.6\times10^4$ | | HSDB (2015) | Q | 99 |
| nithiazide<br>$C_6H_8N_4O_3S$<br>[139-94-6]<br>FQSUTLQHSDLLAN-UHFFFAOYSA-N | $6.2\times10^9$ | | HSDB (2015) | Q | 99 |
| 2-methylbenzenesulfonamide<br>$C_7H_9NO_2S$<br>(o-toluenesulfonamide)<br>[88-19-7]<br>YCMLQMDWSXFTIF-UHFFFAOYSA-N | $2.1\times10^1$ | | HSDB (2015) | Q | 99 |
| 4-methylbenzenesulfonamide<br>$C_7H_9NO_2S$<br>(p-toluenesulfonamide)<br>[70-55-3]<br>LMYRWZFENFIFIT-UHFFFAOYSA-N | $2.1\times10^1$ | | HSDB (2015) | Q | 99 |
| ethidimuron<br>$C_7H_{12}N_4O_3S_2$<br>[30043-49-3]<br>KCOCSOWTADCKOL-UHFFFAOYSA-N | $1.4\times10^8$ | | MacBean (2012a) | ? | |
| oxamyl<br>$C_7H_{13}N_3O_3S$<br>[23135-22-0]<br>KZAUOCCYDRDERY-UITAMQMPSA-N | $4.2\times10^4$<br>$4.2\times10^4$<br>$4.2\times10^4$<br>$3.8\times10^3$<br>$3.8\times10^1$<br>$3.4\times10^7$<br>$6.4\times10^1$<br>$2.0\times10^7$ | | Duchowicz et al. (2020)<br>HSDB (2015)<br>Mackay et al. (2006d)<br>Suntio et al. (1988)<br>Barcelo and Hennion (1997)<br>Duchowicz et al. (2020)<br>Goodarzi et al. (2010)<br>Maniere et al. (2011) | V<br>V<br>V<br>V<br>X<br>Q<br>Q<br>? | 186<br><br><br>12<br>567<br><br>568<br>165 |
| aldicarb<br>$C_7H_{14}N_2O_2S$<br>[116-06-3]<br>QGLZXHRNAYXIBU-WEVVVXLNSA-N | $6.9\times10^3$<br>$6.6\times10^3$<br>$7.9\times10^3$<br>$3.1\times10^3$<br>$3.1\times10^1$<br>$1.9$<br>$8.0\times10^3$<br>$2.6\times10^1$ | | Duchowicz et al. (2020)<br>HSDB (2015)<br>Mackay et al. (2006d)<br>Suntio et al. (1988)<br>Barcelo and Hennion (1997)<br>Suntio et al. (1988)<br>Duchowicz et al. (2020)<br>Goodarzi et al. (2010) | V<br>V<br>V<br>V<br>X<br>C<br>Q<br>Q | 186<br><br><br>12<br>567<br>12<br><br>568, 571 |



Table A9.1: Sulfur (C, H, O, N, Cl, S) (...continued)

| Substance<br>Formula<br>(Trivial Name)<br>[CAS Registry Number]<br>InChIKey | $H_s^{cp}$<br>(at $T^\ominus$)<br>$\left[\dfrac{\text{mol}}{\text{m}^3\,\text{Pa}}\right]$ | $\dfrac{\text{d}\ln H_s^{cp}}{\text{d}(1/T)}$<br><br>[K] | Reference | Type | Note |
|---|---|---|---|---|---|
| aldicarb sulfoxide<br>$C_7H_{14}N_2O_3S$<br>[1646-87-3]<br>BXPMAGSOWXBZHS-UHFFFAOYSA-N | $1.0\times10^4$<br>$4.2\times10^6$ | | Duchowicz et al. (2020)<br>Duchowicz et al. (2020) | V<br>Q | 186 |
| aldicarb sulfone<br>$C_7H_{14}N_2O_4S$<br>[1646-88-4]<br>YRRKLBAKDXSTNC-UHFFFAOYSA-N | $2.9\times10^3$<br>$4.4\times10^6$<br>$3.7\times10^3$ | | Duchowicz et al. (2020)<br>Duchowicz et al. (2020)<br>MacBean (2012a) | V<br>Q<br>? | 186 |
| butocarboxim<br>$C_7H_{14}N_2O_2S$<br>[34681-10-2]<br>SFNPDDSJBGRXLW-UHFFFAOYSA-N | $1.7\times10^4$<br>$1.7\times10^4$ | | HSDB (2015)<br>MacBean (2012a) | V<br>? | |
| butoxycarboxim<br>$C_7H_{14}N_2O_4S$<br>[34681-23-7]<br>CTJBHIROCMPUKL-UHFFFAOYSA-N | $3.5\times10^6$ | | HSDB (2015) | V | |
| saccharin<br>$C_7H_5NO_3S$<br>[81-07-2]<br>CVHZOJJKTDOEJC-UHFFFAOYSA-N | $8.2\times10^3$ | | HSDB (2015) | Q | 99 |
| acibenzolar-S-methyl<br>$C_8H_6N_2OS_2$<br>[135158-54-2]<br>UELITFHSCLAHKR-UHFFFAOYSA-N | $8.3\times10^1$<br>$8.2\times10^1$<br>$1.5\times10^6$<br>$7.7\times10^1$ | | Duchowicz et al. (2020)<br>HSDB (2015)<br>Duchowicz et al. (2020)<br>Maniere et al. (2011) | V<br>V<br>Q<br>? | 186<br><br><br>241, 165 |
| quinomethionate<br>$C_8H_6N_2OS_2$<br>[2439-01-2]<br>FBQQHUGEACOBDN-UHFFFAOYSA-N | $1.6\times10^2$<br>$1.6\times10^2$ | | HSDB (2015)<br>MacBean (2012a) | V<br>? | |
| nifurthiazole<br>$C_8H_6N_4O_4S$<br>[3570-75-0]<br>DUWYZHLZDVCZIO-UHFFFAOYSA-N | $1.3\times10^{12}$ | | HSDB (2015) | Q | 99 |
| 4-methylbenzenesulfonyl isocyanate<br>$C_8H_7NO_3S$<br>[4083-64-1]<br>VLJQDHDVZJXNQL-UHFFFAOYSA-N | $1.7\times10^{-1}$<br><br>$3.2\times10^1$<br>6.7<br>$4.0\times10^4$ | | Zhang et al. (2010)<br><br>Zhang et al. (2010)<br>Zhang et al. (2010)<br>Zhang et al. (2010) | Q<br><br>Q<br>Q<br>Q | 287, 288<br><br>287, 289<br>287, 290<br>287, 291 |
| tinidazole<br>$C_8H_{13}N_3O_4S$<br>[19387-91-8]<br>HJLSLZFTEKNLFI-UHFFFAOYSA-N | $1.9\times10^5$ | | HSDB (2015) | Q | 99 |



Table A9.1: Sulfur (C, H, O, N, Cl, S) (...continued)

| Substance<br>Formula<br>(Trivial Name)<br>[CAS Registry Number]<br>InChIKey | $H_s^{cp}$<br>(at $T^{\ominus}$)<br>$\left[\dfrac{\mathrm{mol}}{\mathrm{m^3\,Pa}}\right]$ | $\dfrac{\mathrm{d}\ln H_s^{cp}}{\mathrm{d}(1/T)}$<br><br>[K] | Reference | Type | Note |
|---|---|---|---|---|---|
| metribuzin<br>$C_8H_{14}N_4OS$<br>[21087-64-9]<br>FOXFZRUHNHCZPX-UHFFFAOYSA-N | $8.2\times10^4$<br>$5.0\times10^4$ | | HSDB (2015)<br>Maniere et al. (2011) | V<br>? | <br>241, 165 |
| tricyclazole<br>$C_9H_7N_3S$<br>[41814-78-2]<br>DQJCHOQLCLEDLL-UHFFFAOYSA-N | $3.2\times10^5$<br>$3.2\times10^5$<br>$5.5\times10^5$ | | Duchowicz et al. (2020)<br>Mackay et al. (2006d)<br>Duchowicz et al. (2020) | V<br>V<br>Q | 186 |
| benzthiazuron<br>$C_9H_9N_3OS$<br>[1929-88-0]<br>DTCJYIIKPVRVDD-UHFFFAOYSA-N | $1.9\times10^6$ | | Ebert et al. (2023) | ? | 365 |
| thidiazuron<br>$C_9H_8N_4OS$<br>[51707-55-2]<br>HFCYZXMHUIHAQI-UHFFFAOYSA-N | $3.0\times10^7$ | | HSDB (2015) | V | |
| sulfathiazole<br>$C_9H_9N_3O_2S_2$<br>[72-14-0]<br>JNMRHUJNCSQMMB-UHFFFAOYSA-N | $1.7\times10^8$ | | HSDB (2015) | Q | 99 |
| sulfamethizole<br>$C_9H_{10}N_4O_2S_2$<br>[144-82-1]<br>VACCAVUAMIDAGB-UHFFFAOYSA-N | $3.8\times10^8$ | | HSDB (2015) | Q | 99 |
| ethiozin<br>$C_9H_{16}N_4OS$<br>[64529-56-2]<br>ADZSGNDOZREKJK-UHFFFAOYSA-N | $2.0\times10^2$ | | MacBean (2012a) | ? | |
| tebuthiuron<br>$C_9H_{16}N_4OS$<br>[34014-18-1]<br>HBPDKDSFLXWOAE-UHFFFAOYSA-N | $6.9\times10^4$<br>$3.2\times10^4$<br>$8.2\times10^4$ | | Keshavarz et al. (2022)<br>Duchowicz et al. (2020)<br>Duchowicz et al. (2020) | Q<br>Q<br>? | <br><br>185, 21 |
| molinate<br>$C_9H_{17}NOS$<br>[2212-67-1]<br>DEDOPGXGGQYYMW-UHFFFAOYSA-N | 7.7<br>2.2<br>1.7<br>6.9<br>$1.1\times10^1$<br>$1.0\times10^1$<br>7.6<br>7.3<br>2.3 | <br><br>7300 | Watanabe (1993)<br>Sagebiel et al. (1992)<br>Sagebiel et al. (1992)<br>Mackay et al. (2006d)<br>Sagebiel et al. (1992)<br>Woodrow et al. (1990)<br>Armbrust (2000)<br>Hilal et al. (2008)<br>Modarresi et al. (2007) | M<br>M<br>M<br>V<br>V<br>V<br>C<br>Q<br>Q | <br>12<br><br><br>12<br><br><br><br>67 |





Table A9.1: Sulfur (C, H, O, N, Cl, S) (. . . continued)

| Substance / Formula / (Trivial Name) / [CAS Registry Number] / InChIKey | $H_s^{cp}$ (at $T^\ominus$) $\left[\dfrac{\text{mol}}{\text{m}^3\,\text{Pa}}\right]$ | $\dfrac{\text{d}\ln H_s^{cp}}{\text{d}(1/T)}$ [K] | Reference | Type | Note |
|---|---|---|---|---|---|
| thiofanox $C_9H_{18}N_2O_2S$ [39196-18-4] FZSVSABTBYGOQH-UHFFFAOYSA-N | $1.1\times10^3$ $3.4\times10^3$ | | Duchowicz et al. (2020) Duchowicz et al. (2020) | V Q | 186 |
| amidosulfuron $C_9H_{15}N_5O_7S_2$ [120923-37-7] CTTHWASMBLQOFR-UHFFFAOYSA-N | $6.4\times10^5$ $1.9\times10^3$ $1.5\times10^5$ | | Maniere et al. (2011) Maniere et al. (2011) Maniere et al. (2011) | ? ? ? | 12, 493, 165 12, 577, 165 12, 573, 165 |
| S-ethyl dipropylthiocarbamate $C_9H_{19}NOS$ (eptam; EPTC) [759-94-4] GUVLYNGULCJVDO-UHFFFAOYSA-N | $5.6\times10^{-1}$ $3.8\times10^{-2}$ $6.2\times10^{-1}$ $9.8\times10^{-1}$ $4.2\times10^{-1}$ $9.8\times10^{-1}$ $7.4\times10^{-1}$ $1.0\times10^{-2}$ $1.1\times10^{-2}$ $8.2\times10^{-1}$ $1.2$ | 9100 4800 4800 4800 | Reyes-Pérez et al. (2008) Breiter et al. (1998) HSDB (2015) Mackay et al. (2006d) Breiter et al. (1998) Suntio et al. (1988) Burkhard and Guth (1981) Barcelo and Hennion (1997) Goodarzi et al. (2010) Hilal et al. (2008) Modarresi et al. (2007) Kühne et al. (2005) Kühne et al. (2005) | M M V V V V V X Q Q Q Q ? | 12 567 568 67 |
| thiabendazole $C_{10}H_7N_3S$ [148-79-8] WJCNZQLZVWNLKY-UHFFFAOYSA-N | $4.7\times10^5$ $4.7\times10^5$ $4.7\times10^5$ $1.3\times10^5$ $2.7\times10^5$ | | Duchowicz et al. (2020) HSDB (2015) Mackay et al. (2006d) Duchowicz et al. (2020) Maniere et al. (2011) | V V V Q ? | 186 241, 165 |
| benzo[$b$]thiophene-4-ol, methylcarbamate $C_{10}H_9NO_2S$ (mobam) [1079-33-0] BOTUVXISJHKZKJ-UHFFFAOYSA-N | $5.8\times10^3$ | | HSDB (2015) | Q | 99 |
| sulfamethoxazole $C_{10}H_{11}N_3O_3S$ [723-46-6] JLKIGFTWXXRPMT-UHFFFAOYSA-N | $1.5\times10^7$ | | HSDB (2015) | Q | 99 |
| bentazone $C_{10}H_{12}N_2O_3S$ [25057-89-0] ZOMSMJKLGFBRBS-UHFFFAOYSA-N | $4.5\times10^3$ $1.3\times10^2$ $3.6\times10^2$ $1.4\times10^4$ $4.7\times10^5$ | | HSDB (2015) Barcelo and Hennion (1997) Goodarzi et al. (2010) Maniere et al. (2011) Maniere et al. (2011) | V X Q ? ? | 567 568, 569 12, 165 12, 165 |



Table A9.1: Sulfur (C, H, O, N, Cl, S) (. . . continued)

| Substance Formula (Trivial Name) [CAS Registry Number] InChIKey | $H_s^{cp}$ (at $T^\ominus$) $\left[\dfrac{\mathrm{mol}}{\mathrm{m^3\,Pa}}\right]$ | $\dfrac{\mathrm{d}\ln H_s^{cp}}{\mathrm{d}(1/T)}$ [K] | Reference | Type | Note |
|---|---|---|---|---|---|
| buthidazole C$_{10}$H$_{16}$N$_4$O$_2$S [55511-98-3] SWMGXKSQWDSBKV-UHFFFAOYSA-N | $4.8\times10^6$ | | MacBean (2012a) | ? | |
| thiodicarb C$_{10}$H$_{18}$N$_4$O$_4$S$_3$ [59669-26-0] XDOTVMNBCQVZKG-UHFFFAOYSA-N | $1.1\times10^1$ $2.3\times10^1$ | | HSDB (2015) Mackay et al. (2006d) | V V | |
| pebulate C$_{10}$H$_{21}$NOS [1114-71-2] SGEJQUSYQTVSIU-UHFFFAOYSA-N | $4.1\times10^{-2}$ $3.8\times10^{-1}$ $8.6\times10^{-2}$ $8.4\times10^{-4}$ $6.6\times10^{-3}$ $6.4\times10^{-1}$ $8.9\times10^{-1}$ $>2.3\times10^{10}$ | | HSDB (2015) Mackay et al. (2006d) Suntio et al. (1988) Barcelo and Hennion (1997) Goodarzi et al. (2010) Hilal et al. (2008) Modarresi et al. (2007) MacBean (2012a) | V V V X Q Q Q ? | 12 567 568 67 |
| vernolate C$_{10}$H$_{21}$NOS [1929-77-7] OKUGPJPKMAEJOE-UHFFFAOYSA-N | $3.2\times10^{-1}$ $4.9\times10^{-1}$ $4.9\times10^{-1}$ $4.8\times10^{-3}$ $2.9\times10^{-3}$ $6.5\times10^{-1}$ $9.0\times10^{-1}$ | | HSDB (2015) Mackay et al. (2006d) Suntio et al. (1988) Barcelo and Hennion (1997) Goodarzi et al. (2010) Hilal et al. (2008) Modarresi et al. (2007) | V V V X Q Q Q | 12 567 568 67 |
| biotin C$_{10}$H$_{16}$N$_2$O$_3$S [58-85-5] YBJHBAHKTGYVGT-ZKWXMUAHSA-N | $5.6\times10^2$ | | Abraham et al. (2019) | Q | |
| methabenzthiazuron C$_{10}$H$_{11}$N$_3$OS [18691-97-9] RRVIAQKBTUQODI-UHFFFAOYSA-N | $1.8\times10^4$ $6.6\times10^4$ | | Duchowicz et al. (2020) Duchowicz et al. (2020) | V Q | 186 |
| sulfisoxazole C$_{11}$H$_{13}$N$_3$O$_3$S [127-69-5] NHUHCSRWZMLRLA-UHFFFAOYSA-N | $6.2\times10^6$ | | HSDB (2015) | Q | 99 |
| ethiofencarb C$_{11}$H$_{15}$NO$_2$S [29973-13-5] HEZNVIYQEUHLNI-UHFFFAOYSA-N | $8.6\times10^3$ $8.2\times10^3$ $7.7\times10^2$ | | Duchowicz et al. (2020) HSDB (2015) Duchowicz et al. (2020) | V V Q | 186 |



Table A9.1: Sulfur (C, H, O, N, Cl, S) (. . . continued)

| Substance Formula (Trivial Name) [CAS Registry Number] InChIKey | $H_s^{cp}$ (at $T^{\ominus}$) $\left[\dfrac{\text{mol}}{\text{m}^3\,\text{Pa}}\right]$ | $\dfrac{\text{d}\ln H_s^{cp}}{\text{d}(1/T)}$ [K] | Reference | Type | Note |
|---|---|---|---|---|---|
| methiocarb $C_{11}H_{15}NO_2S$ [2032-65-7] YFBPRJGDJKVWAH-UHFFFAOYSA-N | 8.3 $8.4\times10^3$ | | Mackay et al. (2006d) MacBean (2012b) | V X | 350 |
| cycloate $C_{11}H_{21}NOS$ [1134-23-2] DFCAFRGABIXSDS-UHFFFAOYSA-N | 1.9 3.7 3.2 | | HSDB (2015) Hilal et al. (2008) Modarresi et al. (2007) | V Q Q | 67 |
| methoprotryn $C_{11}H_{21}N_5OS$ [841-06-5] DDUIUBPJPOKOMV-UHFFFAOYSA-N | $3.1\times10^4$ $1.5\times10^5$ $2.0\times10^5$ $3.1\times10^4$ | | HSDB (2015) Hilal et al. (2008) Abraham et al. (2007) MacBean (2012a) | V Q Q ? | |
| butylate $C_{11}H_{23}NOS$ [2008-41-5] BMTAFVWTTFSTOG-UHFFFAOYSA-N | $1.2\times10^{-1}$ 1.8 $1.8\times10^{-2}$ $2.1\times10^{-1}$ $5.8\times10^{-1}$ $6.0\times10^{-1}$ | | HSDB (2015) Mackay et al. (2006d) Suntio et al. (1988) Barcelo and Hennion (1997) Goodarzi et al. (2010) Hilal et al. (2008) Modarresi et al. (2007) | V V V X Q Q Q | 558 12 567 568 67 |
| 2-octyl-3(2H)-isothiazolone $C_{11}H_{19}NOS$ [26530-20-1] JPMIIZHYYWMHDT-UHFFFAOYSA-N | $4.8\times10^2$ 1.8 | | Duchowicz et al. (2020) Duchowicz et al. (2020) | V Q | 186 |
| metsulfovax $C_{12}H_{12}N_2OS$ [21452-18-6] UDSJPFPDKCMYBD-UHFFFAOYSA-N | $9.2\times10^5$ | | Ebert et al. (2023) | ? | 318 |
| carboxin $C_{12}H_{13}NO_2S$ [5234-68-4] GYSSRZJIHXQEHQ-UHFFFAOYSA-N | $3.1\times10^4$ $3.1\times10^4$ $6.4\times10^4$ $7.6\times10^2$ $3.1\times10^4$ | | Duchowicz et al. (2020) HSDB (2015) Mackay et al. (2006d) Duchowicz et al. (2020) Maniere et al. (2011) | V V V Q ? | 186 241, 165 |
| oxycarboxin $C_{12}H_{13}NO_4S$ [5259-88-1] AMEKQAFGQBKLKX-UHFFFAOYSA-N | $9.3\times10^5$ $9.0\times10^5$ $2.8\times10^3$ $5.3\times10^5$ | | Duchowicz et al. (2020) HSDB (2015) Mackay et al. (2006d) Duchowicz et al. (2020) | V V V Q | 186 |
| thifensulfuron-methyl $C_{12}H_{13}N_5O_6S_2$ [79277-27-3] AHTPATJNIAFOLR-UHFFFAOYSA-N | $3.4\times10^8$ $1.0\times10^{14}$ $3.6\times10^{12}$ | | HSDB (2015) Maniere et al. (2011) Maniere et al. (2011) | V ? ? | 241, 493, 165 241, 570, 165 |



Table A9.1: Sulfur (C, H, O, N, Cl, S) (... continued)

| Substance Formula (Trivial Name) [CAS Registry Number] InChIKey | $H_s^{cp}$ (at $T^{\ominus}$) $\left[\dfrac{\text{mol}}{\text{m}^3\,\text{Pa}}\right]$ | $\dfrac{\text{d}\ln H_s^{cp}}{\text{d}(1/T)}$ [K] | Reference | Type | Note |
|---|---|---|---|---|---|
| sulfamethazine $C_{12}H_{14}N_4O_2S$ [57-68-1] ASWVTGNCAZCNNR-UHFFFAOYSA-N | $3.2\times10^7$ | | HSDB (2015) | Q | 99 |
| thiophanate-methyl $C_{12}H_{14}N_4O_4S_2$ [23564-05-8] QGHREAKMXXNCOA-UHFFFAOYSA-N | $8.2\times10^3$ $7.9\times10^2$ $6.0\times10^3$ | | HSDB (2015) Mackay et al. (2006d) Maniere et al. (2011) | V V ? | 241, 165 |
| 4,4'-oxydi(benzenesulphonohydrazide) $C_{12}H_{14}N_4O_5S_2$ [80-51-3] NBOCQTNZUPTTEI-UHFFFAOYSA-N | $7.8\times10^{11}$ | | HSDB (2015) | Q | 99 |
| albendazole $C_{12}H_{15}N_3O_2S$ [54965-21-8] HXHWSAZORRCQMX-UHFFFAOYSA-N | $1.3\times10^8$ | | HSDB (2015) | Q | 99 |
| oryzalin $C_{12}H_{18}N_4O_6S$ [19044-88-3] UNAHYJYOSSSJHH-UHFFFAOYSA-N | $5.2\times10^3$ $5.3\times10^3$ $3.6\times10^7$ $2.5\times10^8$ $3.0\times10^7$ | | HSDB (2015) Mackay et al. (2006d) Maniere et al. (2011) Maniere et al. (2011) Maniere et al. (2011) | V V ? ? ? | 12, 570, 165 12, 572, 165 12, 493, 165 |
| STK366145 $C_{12}H_{19}N_3O_3S$ (N-(2-ethyl(3-methyl-4-nitrosophenyl)amino)ethyl)-methanesulfonamide) [56046-62-9] XWQURWIJAIIPQP-UHFFFAOYSA-N | $9.9\times10^4$ | | HSDB (2015) | V | |
| isomethiozin $C_{12}H_{20}N_4OS$ [57052-04-7] MZTLOILRKLUURT-UHFFFAOYSA-N | $7.9\times10^2$ | | MacBean (2012a) | ? | |



Table A9.1: Sulfur (C, H, O, N, Cl, S) (...continued)

| Substance<br>Formula<br>(Trivial Name)<br>[CAS Registry Number]<br>InChIKey | $H_s^{cp}$<br>(at $T^\ominus$)<br>$\left[\dfrac{\mathrm{mol}}{\mathrm{m^3\,Pa}}\right]$ | $\dfrac{\mathrm{d}\ln H_s^{cp}}{\mathrm{d}(1/T)}$<br><br>[K] | Reference | Type | Note |
|---|---|---|---|---|---|
| thiencarbazone-methyl | $2.1\times10^{12}$ | | Maniere et al. (2011) | ? | 241, 815, 165 |
| $C_{12}H_{14}N_4O_7S_2$ | $5.0\times10^{12}$ | | Maniere et al. (2011) | ? | 241, 577, 165 |
| [317815-83-1] | $1.3\times10^{13}$ | | Maniere et al. (2011) | ? | 241, 493, 165 |
| XSKZXGDFSCCXQX-UHFFFAOYSA-N | $1.2\times10^{13}$ | | Maniere et al. (2011) | ? | 241, 573, 165 |
| 2,8-dinitrodibenzothiophene<br>$C_{12}H_6N_2O_4S$<br>[109041-38-5]<br>VMQHOWOVMXIROE-UHFFFAOYSA-N | $4.0\times10^{4}$ | | Parnis et al. (2015) | Q | 369 |
| 2-nitrodibenzothiophene<br>$C_{12}H_7NO_2S$<br>[6639-36-7]<br>GXLYVLHWXVRVKI-UHFFFAOYSA-N | $1.2\times10^{2}$ | | Parnis et al. (2015) | Q | 369 |
| azimsulfuron | $2.0\times10^{9}$ | | Maniere et al. (2011) | ? | 241, 493, 165 |
| $C_{13}H_{16}N_{10}O_5S$ | $1.2\times10^{8}$ | | Maniere et al. (2011) | ? | 241, 570, 165 |
| [120162-55-2] | $1.1\times10^{10}$ | | Maniere et al. (2011) | ? | 241, 573, 165 |
| MAHPNPYYQAIOJN-UHFFFAOYSA-N | | | | | |
| fenothiocarb<br>$C_{13}H_{19}NO_2S$<br>[62850-32-2]<br>HMIBKHHNXANVHR-UHFFFAOYSA-N | $5.3\times10^{2}$ | | Ebert et al. (2023) | ? | 316 |
| isobornyl thiocyanoacetate<br>$C_{13}H_{19}NO_2S$<br>[115-31-1]<br>IXEVGHXRXDBAOB-RUETXSTFSA-N | $3.8\times10^{1}$ | | HSDB (2015) | Q | 99 |
| nitralin<br>$C_{13}H_{19}N_3O_6S$<br>[4726-14-1]<br>UMKANAFDOQQUKE-UHFFFAOYSA-N | $1.4\times10^{3}$<br>$7.2\times10^{-3}$<br>$7.2\times10^{-3}$ | | HSDB (2015)<br>Mackay et al. (2006d)<br>Suntio et al. (1988) | V<br>V<br>V | <br><br>12 |
| bupirimate<br>$C_{13}H_{24}N_4O_3S$<br>[41483-43-6]<br>DSKJPMWIHSOYEA-UHFFFAOYSA-N | $6.9\times10^{2}$<br>$1.0\times10^{2}$<br>$1.5\times10^{3}$<br>$7.4\times10^{-1}$ | | Duchowicz et al. (2020)<br>Mackay et al. (2006d)<br>Duchowicz et al. (2020)<br>Maniere et al. (2011) | V<br>V<br>Q<br>? | 186<br><br><br>12, 165 |
| timolol<br>$C_{13}H_{24}N_4O_3S$<br>[26839-75-8]<br>BLJRIMJGRPQVNF-JTQLQIEISA-N | $2.3\times10^{11}$ | | HSDB (2015) | Q | 99 |



Table A9.1: Sulfur (C, H, O, N, Cl, S) (...continued)

| Substance<br>Formula<br>(Trivial Name)<br>[CAS Registry Number]<br>InChIKey | $H_s^{cp}$<br>(at $T^{\ominus}$)<br>$\left[\dfrac{\mathrm{mol}}{\mathrm{m^3\,Pa}}\right]$ | $\dfrac{\mathrm{d}\ln H_s^{cp}}{\mathrm{d}(1/T)}$<br><br>[K] | Reference | Type | Note |
|---|---|---|---|---|---|
| triazamate<br>$C_{13}H_{22}N_4O_3S$<br>[112143-82-5]<br>NKNFWVNSBIXGLL-UHFFFAOYSA-N | $8.6\times10^4$<br>$1.2\times10^5$ | | Duchowicz et al. (2020)<br>Duchowicz et al. (2020) | V<br>Q | 186 |
| dithianone<br>$C_{14}H_4N_2O_2S_2$<br>[3347-22-6]<br>PYZSVQVRHDXQSL-UHFFFAOYSA-N | $1.7\times10^5$<br><br>$7.4\times10^6$ | | HSDB (2015)<br>Mackay et al. (2006d)<br>Maniere et al. (2011) | V<br>V<br>? | <br>558<br>12, 165 |
| N-(cyclohexylthio)phthalimide<br>$C_{14}H_{15}NO_2S$<br>[17796-82-6]<br>UEZWYKZHXASYJN-UHFFFAOYSA-N | $1.5\times10^2$ | | HSDB (2015) | Q | 99 |
| metsulfuron-methyl<br>$C_{14}H_{15}N_5O_6S$<br>[74223-64-6]<br>RSMUVYRMZCOLBH-UHFFFAOYSA-N | $7.5\times10^{10}$<br>$2.1\times10^8$ | | HSDB (2015)<br>Maniere et al. (2011) | V<br>? | <br>241, 165 |
| rimsulfuron<br>$C_{14}H_{17}N_5O_7S_2$<br>[122931-48-0]<br>MEFOUWRMVYJCQC-UHFFFAOYSA-N | $1.5\times10^4$<br>$9.1\times10^6$<br>$2.2\times10^5$<br>$1.2\times10^7$ | | HSDB (2015)<br>Maniere et al. (2011)<br>Maniere et al. (2011)<br>Maniere et al. (2011) | V<br>?<br>?<br>? | <br>573, 165<br>570, 165<br>493, 165 |
| thiophanate<br>$C_{14}H_{18}N_4O_4S_2$<br>[23564-06-9]<br>YFNCATAIYKQPOO-UHFFFAOYSA-N | $1.9\times10^7$ | | HSDB (2015) | Q | 99 |
| sumatriptan<br>$C_{14}H_{21}N_3O_2S$<br>[103628-46-2]<br>KQKPFRSPSRPDEB-UHFFFAOYSA-N | $2.2\times10^8$ | | HSDB (2015) | Q | 99 |
| mesotrione<br>$C_{14}H_{13}NO_7S$<br>[104206-82-8]<br>KPUREKXXPHOJQT-UHFFFAOYSA-N | $>2.0\times10^6$ | | Maniere et al. (2011) | ? | 12, 165 |
| prosulfocarb<br>$C_{14}H_{21}NOS$<br>[52888-80-9]<br>NQLVQOSNDJXLKG-UHFFFAOYSA-N | $7.6\times10^2$<br>$9.5\times10^1$<br>$6.6\times10^1$ | | Duchowicz et al. (2020)<br>Duchowicz et al. (2020)<br>Maniere et al. (2011) | V<br>Q<br>? | 186<br><br>12, 165 |
| pyriftalid<br>$C_{15}H_{14}N_2O_4S$<br>[135186-78-6]<br>RRKHIAYNPVQKEF-UHFFFAOYSA-N | $2.5\times10^5$ | | Ebert et al. (2023) | ? | 318 |



Table A9.1: Sulfur (C, H, O, N, Cl, S) (…continued)

| Substance<br>Formula<br>(Trivial Name)<br>[CAS Registry Number]<br>InChIKey | $H_s^{cp}$<br>(at $T^\ominus$)<br>$\left[\dfrac{\mathrm{mol}}{\mathrm{m}^3\,\mathrm{Pa}}\right]$ | $\dfrac{\mathrm{d}\ln H_s^{cp}}{\mathrm{d}(1/T)}$<br><br>[K] | Reference | Type | Note |
|---|---|---|---|---|---|
| nicosulfuron<br>$C_{15}H_{18}N_6O_6S$<br>[111991-09-4]<br>RTCOGUMHFFWOJV-UHFFFAOYSA-N | $6.8\times10^{10}$ | | Maniere et al. (2011) | ? | 12, 165 |
| sulfometuron methyl<br>$C_{15}H_{16}N_4O_5S$<br>[74222-97-2]<br>ZDXMLEQEMNLCQG-UHFFFAOYSA-N | $8.2\times10^{12}$<br>$1.9\times10^8$ | | Armbrust (2000)<br>HSDB (2015) | C<br>Q | <br>99 |
| tribenuron-methyl<br>$C_{15}H_{17}N_5O_6S$<br>[101200-48-0]<br>VLCQZHSMCYCDJL-UHFFFAOYSA-N | $9.7\times10^7$<br>$6.1\times10^7$<br>$8.4\times10^8$<br>$9.7\times10^7$<br>$1.1\times10^9$<br><br>$>7.5\times10^9$ | | MacBean (2012b)<br>Keshavarz et al. (2022)<br>Duchowicz et al. (2020)<br>Duchowicz et al. (2020)<br>Maniere et al. (2011)<br><br>Maniere et al. (2011) | X<br>Q<br>Q<br>?<br>?<br><br>? | 350<br><br><br>185, 21<br>241, 570,<br>165<br>241, 573,<br>165 |
| ethametsulfuron-methyl<br>$C_{15}H_{18}N_6O_6S$<br>[97780-06-8]<br>ZINJLDJMHCUBIP-UHFFFAOYSA-N | $2.1\times10^9$ | | Ebert et al. (2023) | ? | 318 |
| propoxycarbazone<br>$C_{15}H_{18}N_4O_7S$<br>[145026-81-9]<br>JTHMVYBOQLDDIY-UHFFFAOYSA-N | $7.0\times10^{11}$ | | HSDB (2015) | Q | 99 |
| dimepiperate<br>$C_{15}H_{21}NOS$<br>[61432-55-1]<br>BWUPSGJXXPATLU-UHFFFAOYSA-N | $2.8\times10^2$ | | Ebert et al. (2023) | ? | 318 |
| esprocarb<br>$C_{15}H_{23}NOS$<br>[85785-20-2]<br>BXEHUCNTIZGSOJ-UHFFFAOYSA-N | $1.8\times10^1$<br>$1.5\times10^1$ | | Duchowicz et al. (2020)<br>Duchowicz et al. (2020) | V<br>Q | 186<br> |
| valdecoxib<br>$C_{16}H_{14}N_2O_3S$<br>[181695-72-7]<br>LNPDTQAFDNKSHK-UHFFFAOYSA-N | $4.5\times10^5$ | | HSDB (2015) | Q | 99 |
| topramezone<br>$C_{16}H_{17}N_3O_5S$<br>[210631-68-8]<br>BPPVUXSMLBXYGG-UHFFFAOYSA-N | $1.0\times10^{12}$ | | HSDB (2015) | Q | 99 |





Table A9.1: Sulfur (C, H, O, N, Cl, S) (...continued)

| Substance Formula (Trivial Name) [CAS Registry Number] InChIKey | $H_s^{cp}$ (at $T^{\ominus}$) $\left[\dfrac{\text{mol}}{\text{m}^3\,\text{Pa}}\right]$ | $\dfrac{\text{d}\ln H_s^{cp}}{\text{d}(1/T)}$ [K] | Reference | Type | Note |
|---|---|---|---|---|---|
| sulfosulfuron | $4.3\times10^5$ | | HSDB (2015) | V | |
| $C_{16}H_{18}N_6O_7S_2$ | $3.4\times10^7$ | | Maniere et al. (2011) | ? | 12, 573, 165 |
| [141776-32-1] | $1.2\times10^6$ | | Maniere et al. (2011) | ? | 12, 570, 165 |
| RBSXHDIPCIWOMG-UHFFFAOYSA-N | $1.1\times10^8$ | | Maniere et al. (2011) | ? | 12, 493, 165 |
| orthosulfamuron | $1.3\times10^4$ | | HSDB (2015) | V | |
| $C_{16}H_{20}N_6O_6S$ | | | | | |
| [213464-77-8] | | | | | |
| UCDPMNSCCRBWIC-UHFFFAOYSA-N | | | | | |
| cafenstrole | $8.0\times10^5$ | | Ebert et al. (2023) | ? | 318 |
| $C_{16}H_{22}N_4O_3S$ | | | | | |
| [125306-83-4] | | | | | |
| HFEJHAAIJZXXRE-UHFFFAOYSA-N | | | | | |
| buprofezin | 2.4 | | Duchowicz et al. (2020) | V | 186 |
| $C_{16}H_{23}N_3OS$ | 2.3 | | HSDB (2015) | V | |
| [69327-76-0] | $2.5\times10^2$ | | Duchowicz et al. (2020) | Q | |
| PRLVTUNWOQKEAI-UHFFFAOYSA-N | | | | | |
| bensulfuron methyl | $7.0\times10^{10}$ | | Armbrust (2000) | C | |
| $C_{16}H_{18}N_4O_7S$ | $2.5\times10^{12}$ | | Maniere et al. (2011) | ? | 573, 165 |
| [83055-99-6] | $5.0\times10^{10}$ | | Maniere et al. (2011) | ? | 493, 165 |
| XMQFTWRPUQYINF-UHFFFAOYSA-N | $1.7\times10^9$ | | Maniere et al. (2011) | ? | 570, 165 |
| mefenacet | $2.1\times10^4$ | | Duchowicz et al. (2020) | V | 186 |
| $C_{16}H_{14}N_2O_2S$ | $3.4\times10^4$ | | Duchowicz et al. (2020) | Q | |
| [73250-68-7] | | | | | |
| XIGAUIHYSDTJHW-UHFFFAOYSA-N | | | | | |
| alanycarb | $1.1\times10^4$ | | Duchowicz et al. (2020) | V | 186 |
| $C_{17}H_{25}N_3O_4S_2$ | $1.8\times10^5$ | | Duchowicz et al. (2020) | Q | |
| [83130-01-2] | | | | | |
| GMAUQNJOSOMMHI-JXAWBTAJSA-N | | | | | |
| fenamidone | $8.6\times10^4$ | | Ebert et al. (2023) | ? | 365 |
| $C_{17}H_{17}N_3OS$ | | | | | |
| [161326-34-7] | | | | | |
| LMVPQMGRYSRMIW-KRWDZBQOSA-N | | | | | |
| esomeprazole | $3.3\times10^{13}$ | | HSDB (2015) | Q | 99 |
| $C_{17}H_{19}N_3O_3S$ | | | | | |
| [119141-88-7] | | | | | |
| SUBDBMMJDZJVOS-UHFFFAOYSA-N | | | | | |



Table A9.1: Sulfur (C, H, O, N, Cl, S) (... continued)

| Substance Formula (Trivial Name) [CAS Registry Number] InChIKey | $H_s^{cp}$ (at $T^\ominus$) $\left[\dfrac{\mathrm{mol}}{\mathrm{m}^3\,\mathrm{Pa}}\right]$ | $\dfrac{\mathrm{d}\ln H_s^{cp}}{\mathrm{d}(1/T)}$ [K] | Reference | Type | Note |
|---|---|---|---|---|---|
| foramsulfuron $C_{17}H_{20}N_6O_7S$ [173159-57-4] PXDNXJSDGQBLKS-UHFFFAOYSA-N | $1.7\times10^{11}$ $1.7\times10^{11}$ | | HSDB (2015) Maniere et al. (2011) | V ? | 12, 165 |
| sethoxydim $C_{17}H_{29}NO_3S$ [74051-80-2] NMHGOXYVOKDNHF-UHFFFAOYSA-N | $4.5\times10^5$ | | HSDB (2015) | Q | 99 |
| mesosulfuron-methyl $C_{17}H_{21}N_5O_9S_2$ [208465-21-8] NIFKBBMCXCMCAO-UHFFFAOYSA-N | $9.0\times10^{10}$ $8.7\times10^{12}$ $2.7\times10^{11}$ $4.1\times10^9$ | | HSDB (2015) Maniere et al. (2011) Maniere et al. (2011) Maniere et al. (2011) | V ? ? ? | 12, 573, 165 12, 493, 165 12, 570, 165 |
| cycloxydim $C_{17}H_{27}NO_3S$ [101205-02-1] HAHCNFVGRVWFIP-VKAVYKQESA-N | $1.6\times10^4$ | | Maniere et al. (2011) | ? | 12, 165 |
| fenpyrazamine $C_{17}H_{21}N_3O_2S$ [473798-59-3] UTOHZQYBSYOOGC-UHFFFAOYSA-N | $6.2\times10^3$ | | Maniere et al. (2011) | ? | 12, 165 |
| cyprosulfamide $C_{18}H_{18}N_2O_5S$ [221667-31-8] OAWUUPVZMNKZRY-UHFFFAOYSA-N | $7.0\times10^6$ $9.1\times10^2$ $2.9\times10^5$ | | Maniere et al. (2011) Maniere et al. (2011) Maniere et al. (2011) | ? ? ? | 241, 573, 165 241, 577, 165 241, 493, 165 |
| rosiglitazone $C_{18}H_{19}N_3O_3S$ [122320-73-4] YASAKCUCGLMORW-UHFFFAOYSA-N | $5.8\times10^8$ | | HSDB (2015) | Q | 99 |
| rabeprazole $C_{18}H_{21}N_3O_3S$ [117976-89-3] YREYEVIYCVEVJK-UHFFFAOYSA-N | $8.2\times10^{11}$ | | HSDB (2015) | Q | 99 |
| pyributicarb $C_{18}H_{22}N_2O_2S$ [88678-67-5] VTRWMTJQBQJKQH-UHFFFAOYSA-N | $4.3\times10^1$ | | Ebert et al. (2023) | ? | 318 |





Table A9.1: Sulfur (C, H, O, N, Cl, S) (. . . continued)

| Substance Formula (Trivial Name) [CAS Registry Number] InChIKey | $H_s^{cp}$ (at $T^\ominus$) $\left[\dfrac{\mathrm{mol}}{\mathrm{m^3\,Pa}}\right]$ | $\dfrac{\mathrm{d}\ln H_s^{cp}}{\mathrm{d}(1/T)}$ [K] | Reference | Type | Note |
|---|---|---|---|---|---|
| furathiocarb $C_{18}H_{26}N_2O_5S$ [65907-30-4] HAWJXYBZNNRMNO-UHFFFAOYSA-N | $7.6\times10^3$ | | HSDB (2015) | V | |
| lincomycin $C_{18}H_{34}N_2O_6S$ [154-21-2] OJMMVQQUTAEWLP-UHFFFAOYSA-N | $3.3\times10^{17}$ | | HSDB (2015) | Q | 99 |
| pioglitazone $C_{19}H_{20}N_2O_3S$ [111025-46-8] HYAFETHFCAUJAY-UHFFFAOYSA-N | $5.8\times10^6$ | | HSDB (2015) | Q | 99 |
| isofetamid $C_{20}H_{25}NO_3S$ [875915-78-9] WMKZDPFZIZQROT-UHFFFAOYSA-N | $3.5\times10^4$ | | Ebert et al. (2023) | ? | 318 |
| tamsulosin $C_{20}H_{28}N_2O_5S$ [106133-20-4] DRHKJLXJIQTDTD-OAHLLOKOSA-N | $2.0\times10^9$ | | HSDB (2015) | Q | 99 |
| carbosulfan $C_{20}H_{32}N_2O_3S$ [55285-14-8] JLQUFIHWVLZVTJ-UHFFFAOYSA-N | $1.9\times10^1$ 1.7 | | Duchowicz et al. (2020) Duchowicz et al. (2020) | V Q | 186 |
| sufentanil $C_{22}H_{30}N_2O_2S$ [56030-54-7] GGCSSNBKKAUURC-UHFFFAOYSA-N | $2.4\times10^9$ | | HSDB (2015) | Q | 99 |
| sildenafil $C_{22}H_{30}N_6O_4S$ [139755-83-2] BNRNXUUZRGQAQC-UHFFFAOYSA-N | $1.4\times10^{15}$ | | HSDB (2015) | Q | 99 |
| benfuracarb $C_{20}H_{30}N_2O_5S$ [82560-54-1] FYZBOYWSHKHDMT-UHFFFAOYSA-N | $4.9\times10^2$ | | Ebert et al. (2023) | ? | 318 |
| tirofiban $C_{22}H_{36}N_2O_5S$ [144494-65-5] COKMIXFXJJXBQG-NRFANRHFSA-N | $1.3\times10^9$ | | HSDB (2015) | Q | 99 |





Table A9.1: Sulfur (C, H, O, N, Cl, S) (... continued)

| Substance<br>Formula<br>(Trivial Name)<br>[CAS Registry Number]<br>InChIKey | $H_s^{cp}$<br>(at $T^{\ominus}$)<br>$\left[\dfrac{\text{mol}}{\text{m}^3\,\text{Pa}}\right]$ | $\dfrac{\text{d}\ln H_s^{cp}}{\text{d}(1/T)}$<br><br>[K] | Reference | Type | Note |
|---|---|---|---|---|---|
| vardenafil<br>$C_{23}H_{32}N_6O_4S$<br>[224785-90-4]<br>SECKRCOLJRRGGV-UHFFFAOYSA-N | $5.2\times10^{15}$ | | HSDB (2015) | Q | 99 |
| taurocholic acid<br>$C_{26}H_{45}NO_7S$<br>[81-24-3]<br>WBWWGRHZICKQGZ-HZAMXZRMSA-N | $1.9\times10^{15}$ | | HSDB (2015) | Q | 447 |
| dalfopristin<br>$C_{34}H_{50}N_4O_9S$<br>[112362-50-2]<br>SUYRLXYYZQTJHF-FUODUHIRSA-N | $2.2\times10^{24}$ | | HSDB (2015) | Q | 99 |
| C.I. acid green 3<br>$C_{37}H_{37}N_2O_6S_2$<br>[4680-78-8]<br>SRRJCDUOSQWHGS-UHFFFAOYSA-O | $2.0\times10^{23}$ | | HSDB (2015) | Q | 447 |
| tinopal<br>$C_{40}H_{40}N_{12}O_8S_2$<br>[24231-46-7]<br>YGUMVDWOQQJBGA-VAWYXSNFSA-N | $1.2\times10^{38}$<br>$1.4\times10^{40}$<br>$4.2\times10^{26}$<br>$2.2\times10^{37}$ | | Zhang et al. (2010)<br>Zhang et al. (2010)<br>Zhang et al. (2010)<br>Zhang et al. (2010) | Q<br>Q<br>Q<br>Q | 287, 288<br>287, 289<br>287, 290<br>287, 291 |
| quinupristin<br>$C_{53}H_{67}N_9O_{10}S$<br>[120138-50-3]<br>WTHRRGMBUAHGNI-UHFFFAOYSA-N | $4.9\times10^{22}$ | | HSDB (2015) | Q | 99 |
| 3,3,4,4,4-pentafluorobutane-1-thiol<br>$C_4H_5F_5S$<br>[68140-18-1]<br>WEILNYJKAUGBAU-UHFFFAOYSA-N | $5.2\times10^{-5}$<br>$1.4\times10^{-3}$<br>$1.5\times10^{-3}$<br>$1.2\times10^{-5}$ | | Zhang et al. (2010)<br>Zhang et al. (2010)<br>Zhang et al. (2010)<br>Zhang et al. (2010) | Q<br>Q<br>Q<br>Q | 287, 288<br>287, 289<br>287, 290<br>287, 291 |
| 3,3,4,4,5,5,6,6,6-nonafluoro-1-hexanethiol<br>$C_6H_5F_9S$<br>[68140-20-5]<br>GQJXVHYUQPXZOL-UHFFFAOYSA-N | $1.9\times10^{-6}$<br><br>$4.7\times10^{-4}$<br>$3.1\times10^{-4}$<br>$1.9\times10^{-6}$ | | Zhang et al. (2010)<br><br>Zhang et al. (2010)<br>Zhang et al. (2010)<br>Zhang et al. (2010) | Q<br><br>Q<br>Q<br>Q | 287, 288<br><br>287, 289<br>287, 290<br>287, 291 |
| 3,3,4,4,5,5,7,7,8,8,9,9,10,10,10-pentadecafluoro-1-decanethiol<br>$C_{10}H_7F_{15}S$<br>[68140-21-6]<br>ROKKEEDUUQVHFZ-UHFFFAOYSA-N | $9.7\times10^{-9}$<br><br>$6.5\times10^{-6}$<br>$8.6\times10^{-4}$<br>$1.3\times10^{-8}$ | | Zhang et al. (2010)<br><br>Zhang et al. (2010)<br>Zhang et al. (2010)<br>Zhang et al. (2010) | Q<br><br>Q<br>Q<br>Q | 287, 288<br><br>287, 289<br>287, 290<br>287, 291 |



Table A9.1: Sulfur (C, H, O, N, Cl, S) (. . . continued)

| Substance Formula (Trivial Name) [CAS Registry Number] InChIKey | $H_s^{cp}$ (at $T^{\ominus}$) $\left[\dfrac{\text{mol}}{\text{m}^3\,\text{Pa}}\right]$ | $\dfrac{\text{d}\ln H_s^{cp}}{\text{d}(1/T)}$ [K] | Reference | Type | Note |
|---|---|---|---|---|---|
| 3,3,4,4,5,5-hexafluoro-1-(3,3,4,4,5,5-hexafluorohexyldisulfanyl)hexane | $1.2\times10^{-7}$ | | Zhang et al. (2010) | Q | 287, 288 |
| $C_{12}H_{14}F_{12}S_2$ | $9.0\times10^{-6}$ | | Zhang et al. (2010) | Q | 287, 289 |
| [118400-71-8] | $1.9\times10^{-2}$ | | Zhang et al. (2010) | Q | 287, 290 |
| CIZUOSOWGOENRE-UHFFFAOYSA-N | $3.5\times10^{-7}$ | | Zhang et al. (2010) | Q | 287, 291 |
| methanesulfonyl fluoride $CH_3FO_2S$ [558-25-8] KNWQLFOXPQZGPX-UHFFFAOYSA-N | $1.6\times10^{-1}$ | | Ebert et al. (2023) | ? | 318 |
| perfluorobutane sulfonic acid $C_4HF_9O_3S$ (PFBS) [375-73-5] JGTNAGYHADQMCM-UHFFFAOYSA-N | 2.0 | | Plassmann et al. (2011) | E | |
| 4-(pentafluorosulfanyl)phenol $C_6H_5F_5OS$ [774-94-7] XHJLGVIUMCBMHL-UHFFFAOYSA-N | $4.0\times10^{1}$ | | Ebert et al. (2023) | ? | 371 |
| 1H,1H,2H,2H-perfluorohexane sulfonic acid $C_6H_5F_9O_3S$ (4:2 FTS) [757124-72-4] TXGIGTRUEITPSC-UHFFFAOYSA-N | $4.5\times10^{-3}$ | | Abusallout et al. (2022) | M | |
| perfluorohexane sulfonic acid $C_6HF_{13}O_3S$ (PFHxS) [355-46-4] QZHDEAJFRJCDMF-UHFFFAOYSA-N | $5.1\times10^{-1}$ | | Plassmann et al. (2011) | E | |
| 1H,1H,2H,2H-perfluorooctane sulfonic acid $C_8H_5F_{13}O_3S$ (6:2 FTS) [27619-97-2] VIONGDJUYAYOPU-UHFFFAOYSA-N | $2.1\times10^{-3}$ | 11000 | Abusallout et al. (2022) | M | |
| perfluorooctane sulfonic acid $C_8HF_{17}O_3S$ (PFOS) [1763-23-1] YFSUTJLHUFNCNZ-UHFFFAOYSA-N | $9.0\times10^{-4}$ | | Zhang et al. (2010) | Q | 287, 288 |
| | $8.6\times10^{-3}$ | | Zhang et al. (2010) | Q | 287, 289 |
| | $1.6\times10^{-1}$ | | Zhang et al. (2010) | Q | 287, 290 |
| | $9.9\times10^{-3}$ | | Zhang et al. (2010) | Q | 287, 291 |
| | $1.0\times10^{-1}$ | | Arp et al. (2006) | Q | 633 |
| | $4.6\times10^{-3}$ | | Arp et al. (2006) | Q | 634 |



Table A9.1: Sulfur (C, H, O, N, Cl, S) (...continued)

| Substance Formula (Trivial Name) [CAS Registry Number] InChIKey | $H_s^{cp}$ (at $T^\ominus$) $\left[\dfrac{\text{mol}}{\text{m}^3\,\text{Pa}}\right]$ | $\dfrac{\text{d}\ln H_s^{cp}}{\text{d}(1/T)}$ [K] | Reference | Type | Note |
|---|---|---|---|---|---|
| heptadecafluorooctanesulphonyl fluoride $C_8F_{18}O_2S$ (perfluorooctylsulfonyl fluoride) [307-35-7] BHFJBHMTEDLICO-UHFFFAOYSA-N | $1.5\times10^{-7}$ | | HSDB (2015) | Q | 99 |
| 1H,1H,2H,2H-perfluorodecane sulfonic acid $C_{10}H_5F_{17}O_3S$ (8:2 FTS) [39108-34-4] ALVYVCQIFHTIRD-UHFFFAOYSA-N | $2.2\times10^{-3}$ | | Abusallout et al. (2022) | M | |
| fluticasone $C_{22}H_{27}F_3O_4S$ [90566-53-3] MGNNYOODZCAHBA-GQKYHHCASA-N | $4.3\times10^{3}$ | | HSDB (2015) | Q | 99 |
| 3-aminophenylsulfur pentafluoride $C_6H_6F_5NS$ [2993-22-8] NUFLICUHOXHWER-UHFFFAOYSA-N | 6.4 | | Ebert et al. (2023) | ? | 371 |
| 4-aminophenylsulfur pentafluoride $C_6H_6F_5NS$ [2993-24-0] MZGZUHNSMNNSRJ-UHFFFAOYSA-N | 4.0 | | Ebert et al. (2023) | ? | 371 |
| flubenzimine $C_{17}H_{10}F_6N_4S$ [37893-02-0] IZFZCMFMJKDHJZ-UHFFFAOYSA-N | $>2.3\times10^{10}$ | | MacBean (2012a) | ? | |
| thiazafluron $C_6H_7F_3N_4OS$ [25366-23-8] BBJPZPLAZVZTGR-UHFFFAOYSA-N | $3.2\times10^{4}$ | | MacBean (2012a) | ? | |
| 4-nitrophenylsulfur pentafluoride $C_6H_4F_5NO_2S$ [2613-27-6] AGNCKMHGYZKMLN-UHFFFAOYSA-N | $1.6\times10^{-1}$ | | Ebert et al. (2023) | ? | 371 |
| undecafluoro-N-methyl-1-pentanesulfonamide $C_6H_4F_{11}NO_2S$ [68298-13-5] BKKNDZBSSSAGIB-UHFFFAOYSA-N | $3.5\times10^{-4}$ $4.4\times10^{-2}$ $5.6\times10^{-4}$ $6.2\times10^{-1}$ | | Zhang et al. (2010) Zhang et al. (2010) Zhang et al. (2010) Zhang et al. (2010) | Q Q Q Q | 287, 288 287, 289 287, 290 287, 291 |



Table A9.1: Sulfur (C, H, O, N, Cl, S) (... continued)

| Substance Formula (Trivial Name) [CAS Registry Number] InChIKey | $H_s^{cp}$ (at $T^\ominus$) $\left[\dfrac{\text{mol}}{\text{m}^3\,\text{Pa}}\right]$ | $\dfrac{\text{d}\ln H_s^{cp}}{\text{d}(1/T)}$ [K] | Reference | Type | Note |
|---|---|---|---|---|---|
| 1,1,2,2,3,3,4,4,4-nonafluoro-N-(2-hydroxyethyl)-N-methylbutane-1-sulfonamide | $1.8\times10^1$ | | Zhang et al. (2010) | Q | 287, 288 |
| $C_7H_8F_9NO_3S$ | $1.1\times10^1$ | | Zhang et al. (2010) | Q | 287, 289 |
| [34454-97-2] | $4.6\times10^{-1}$ | | Zhang et al. (2010) | Q | 287, 290 |
| DSRUAYIFDCHEEV-UHFFFAOYSA-N | $2.7\times10^2$ | | Zhang et al. (2010) | Q | 287, 291 |
| tridecafluoro-N-methyl-1-hexanesulfonamide | $6.7\times10^{-5}$ | | Zhang et al. (2010) | Q | 287, 288 |
| $C_7H_4F_{13}NO_2S$ | $9.2\times10^{-3}$ | | Zhang et al. (2010) | Q | 287, 289 |
| [68259-15-4] | $2.5\times10^{-4}$ | | Zhang et al. (2010) | Q | 287, 290 |
| HPPDPHZGXWMRHN-UHFFFAOYSA-N | $1.2\times10^{-1}$ | | Zhang et al. (2010) | Q | 287, 291 |
| perfluorooctane sulfonamide | $5.5\times10^{-6}$ | | HSDB (2015) | Q | 99 |
| $C_8H_2F_{17}NO_2S$ | $3.4$ | | Arp et al. (2006) | Q | 633 |
| (PFOSA) | $7.9\times10^{-6}$ | | Arp et al. (2006) | Q | 634 |
| [754-91-6] | | | | | |
| RRRXPPIDPYTNJG-UHFFFAOYSA-N | | | | | |
| emtricitabine | $9.0\times10^{11}$ | | HSDB (2015) | Q | 99 |
| $C_8H_{10}FN_3O_3S$ | | | | | |
| [143491-57-0] | | | | | |
| XQSPYNMVSIKCOC-NTSWFWBYSA-N | | | | | |
| N-ethyl-1,1,2,2,3,3,4,4,4-nonafluoro-N-(2-hydroxyethyl)butane-1-sulfonamide | $1.3\times10^1$ | | Zhang et al. (2010) | Q | 287, 288 |
| $C_8H_{10}F_9NO_3S$ | $8.8$ | | Zhang et al. (2010) | Q | 287, 289 |
| [34449-89-3] | $1.4\times10^{-1}$ | | Zhang et al. (2010) | Q | 287, 290 |
| ZSBOIPHQFKYRMG-UHFFFAOYSA-N | $2.1\times10^2$ | | Zhang et al. (2010) | Q | 287, 291 |
| 1,1,2,2,3,3,4,4,5,5,6,6,7,7,7-pentadecafluoro-N-methylheptane-1-sulphonamide | $1.3\times10^{-5}$ | | Zhang et al. (2010) | Q | 287, 288 |
| $C_8H_4NO_2F_{15}S$ | $1.6\times10^{-3}$ | | Zhang et al. (2010) | Q | 287, 289 |
| [68259-14-3] | $1.2\times10^{-4}$ | | Zhang et al. (2010) | Q | 287, 290 |
| KDHCALLFPWZTPN-UHFFFAOYSA-N | $2.4\times10^{-2}$ | | Zhang et al. (2010) | Q | 287, 291 |
| 1,1,2,2,3,3,4,4,5,5,5-undecafluoro-N-(2-hydroxyethyl)-N-methylpentane-1-sulphonamide | $3.4$ | | Zhang et al. (2010) | Q | 287, 288 |
| $C_8H_8NO_3F_{11}S$ | $2.9$ | | Zhang et al. (2010) | Q | 287, 289 |
| [68555-74-8] | $2.1\times10^{-1}$ | | Zhang et al. (2010) | Q | 287, 290 |
| BRBCKWCOTRPYGH-UHFFFAOYSA-N | $5.6\times10^1$ | | Zhang et al. (2010) | Q | 287, 291 |



Table A9.1: Sulfur (C, H, O, N, Cl, S) (... continued)

| Substance<br>Formula<br>(Trivial Name)<br>[CAS Registry Number]<br>InChIKey | $H_s^{cp}$<br>(at $T^{\ominus}$)<br>$\left[\dfrac{\text{mol}}{\text{m}^3\,\text{Pa}}\right]$ | $\dfrac{\text{d}\ln H_s^{cp}}{\text{d}(1/T)}$<br><br>[K] | Reference | Type | Note |
|---|---|---|---|---|---|
| N-(3-(dimethylamino)propyl)-<br>nonafluoro-1-butanesulfonamide | 2.1 | | Zhang et al. (2010) | Q | 287, 288 |
| $C_9H_{13}F_9N_2O_2S$ | $3.1\times10^1$ | | Zhang et al. (2010) | Q | 287, 289 |
| [68555-77-1] | 1.1 | | Zhang et al. (2010) | Q | 287, 290 |
| XMRMVBVJGSKMEN-UHFFFAOYSA-N | $6.0\times10^2$ | | Zhang et al. (2010) | Q | 287, 291 |
| N-<br>methylperfluorooctanesulphonamide | $4.4\times10^{-4}$ | | Abusallout et al. (2022) | M | |
| $C_9H_4F_{17}NO_2S$ | $2.4\times10^{-6}$ | | Zhang et al. (2010) | Q | 287, 288 |
| (N-MeFOSA) | $2.1\times10^{-4}$ | | Zhang et al. (2010) | Q | 287, 289 |
| [31506-32-8] | $5.2\times10^{-5}$ | | Zhang et al. (2010) | Q | 287, 290 |
| SRMWNTGHXHOWBT-UHFFFAOYSA-N | $5.0\times10^{-3}$ | | Zhang et al. (2010) | Q | 287, 291 |
| N-ethyl-1,1,2,2,3,3,4,4,5,5,5-<br>undecafluoro-N-(2-hydroxyethyl)-<br>1-pentanesulfonamide | 2.5 | | Zhang et al. (2010) | Q | 287, 288 |
| $C_9H_{10}NO_3F_{11}S$ | 2.3 | | Zhang et al. (2010) | Q | 287, 289 |
| [68555-72-6] | $6.0\times10^{-2}$ | | Zhang et al. (2010) | Q | 287, 290 |
| GBPAQIZWHVCENQ-UHFFFAOYSA-N | $4.1\times10^1$ | | Zhang et al. (2010) | Q | 287, 291 |
| 1,1,2,2,3,3,4,4,5,5,6,6,6-<br>tridecafluoro-N-(2-hydroxyethyl)-<br>N-methyl-1-hexanesulfonamide | $6.4\times10^{-1}$ | | Zhang et al. (2010) | Q | 287, 288 |
| $C_9H_8NO_3F_{13}S$ | $6.0\times10^{-1}$ | | Zhang et al. (2010) | Q | 287, 289 |
| [68555-75-9] | $9.2\times10^{-2}$ | | Zhang et al. (2010) | Q | 287, 290 |
| UYIBZOUSVFOPJK-UHFFFAOYSA-N | 9.9 | | Zhang et al. (2010) | Q | 287, 291 |
| N-ethyl-1,1,2,2,3,3,4,4,5,5,6,6,6-<br>tridecafluoro-N-(2-<br>hydroxyethyl)hexane-1-<br>sulfonamide | $4.7\times10^{-1}$ | | Zhang et al. (2010) | Q | 287, 288 |
| $C_{10}H_{10}F_{13}NO_3S$ | $4.6\times10^{-1}$ | | Zhang et al. (2010) | Q | 287, 289 |
| [34455-03-3] | $3.1\times10^{-2}$ | | Zhang et al. (2010) | Q | 287, 290 |
| SSGYCIQAXNQIBC-UHFFFAOYSA-N | 8.8 | | Zhang et al. (2010) | Q | 287, 291 |
| 2-<br>methyl[(nonafluorobutyl)sulfonyl]<br>aminoethyl<br>acrylate | $5.1\times10^{-1}$ | | Zhang et al. (2010) | Q | 287, 288 |
| $C_{10}H_{10}F_9NO_4S$ | $5.3\times10^{-1}$ | | Zhang et al. (2010) | Q | 287, 289 |
| [67584-55-8] | $8.2\times10^1$ | | Zhang et al. (2010) | Q | 287, 290 |
| KEMCLRGAIUJRAN-UHFFFAOYSA-N | $4.8\times10^1$ | | Zhang et al. (2010) | Q | 287, 291 |



Table A9.1: Sulfur (C, H, O, N, Cl, S) (. . . continued)

| Substance<br>Formula<br>(Trivial Name)<br>[CAS Registry Number]<br>InChIKey | $H_s^{cp}$<br>(at $T^\ominus$)<br>$\left[\dfrac{\text{mol}}{\text{m}^3\,\text{Pa}}\right]$ | $\dfrac{\text{d}\ln H_s^{cp}}{\text{d}(1/T)}$<br><br>[K] | Reference | Type | Note |
|---|---|---|---|---|---|
| N-ethylperfluorooctanesulfonamide | $9.4\times10^{-4}$ | | Abusallout et al. (2022) | M | |
| $C_{10}H_6F_{17}NO_2S$ | $1.8\times10^{-6}$ | | HSDB (2015) | Q | 99 |
| (N-EtFOSA) | $1.8\times10^{-6}$ | | Zhang et al. (2010) | Q | 287, 288 |
| [4151-50-2] | $1.4\times10^{-4}$ | | Zhang et al. (2010) | Q | 287, 289 |
| CCEKAJIANROZEO-UHFFFAOYSA-N | $9.5\times10^{-6}$ | | Zhang et al. (2010) | Q | 287, 290 |
| | $3.8\times10^{-3}$ | | Zhang et al. (2010) | Q | 287, 291 |
| | $6.4\times10^{-3}$ | | Arp et al. (2006) | Q | 633 |
| | $7.5\times10^{-3}$ | | Arp et al. (2006) | Q | 634 |
| 1,1,2,2,3,3,4,4,5,5,6,6,7,7,7-pentadecafluoro-N-(2-hydroxyethyl)-N-methylheptane-1-sulphonamide | $1.2\times10^{-1}$ | | Zhang et al. (2010) | Q | 287, 288 |
| $C_{10}H_8NO_3F_{15}S$ | $9.5\times10^{-2}$ | | Zhang et al. (2010) | Q | 287, 289 |
| [68555-76-0] | $4.5\times10^{-2}$ | | Zhang et al. (2010) | Q | 287, 290 |
| UIZUTEDYGNRRNSW-UHFFFAOYSA-N | 2.1 | | Zhang et al. (2010) | Q | 287, 291 |
| N-methyl perfluorooctane sulfonamidoethanol | $2.3\times10^{-2}$ | | Zhang et al. (2010) | Q | 287, 288 |
| $C_{11}H_8F_{17}NO_3S$ | $1.2\times10^{-2}$ | | Zhang et al. (2010) | Q | 287, 289 |
| (MeFOSE) | $1.9\times10^{-2}$ | | Zhang et al. (2010) | Q | 287, 290 |
| [24448-09-7] | $4.3\times10^{-1}$ | | Zhang et al. (2010) | Q | 287, 291 |
| PLGACQRCZCVKGK-UHFFFAOYSA-N | $4.8\times10^{-1}$ | | Arp et al. (2006) | Q | 633 |
| | $2.1\times10^{-3}$ | | Arp et al. (2006) | Q | 634 |
| N-ethyl-1,1,2,2,3,3,4,4,5,5,6,6,7,7,7-pentadecafluoro-N-(2-hydroxyethyl)-1-heptanesulfonamide | $9.0\times10^{-2}$ | | Zhang et al. (2010) | Q | 287, 288 |
| $C_{11}H_{10}NO_3F_{15}S$ | $7.2\times10^{-2}$ | | Zhang et al. (2010) | Q | 287, 289 |
| [68555-73-7] | $1.5\times10^{-2}$ | | Zhang et al. (2010) | Q | 287, 290 |
| HINASMOVWHGCAK-UHFFFAOYSA-N | 1.8 | | Zhang et al. (2010) | Q | 287, 291 |
| mefluidide<br>$C_{11}H_{13}F_3N_2O_3S$<br>[53780-34-0]<br>OKIBNKKYNPBDRS-UHFFFAOYSA-N | $7.6\times10^{5}$ | | HSDB (2015) | Q | 99 |
| N-(3-(dimethylamino)propyl)-1,1,2,2,3,3,4,4,5,5,6,6,6-tridecafluoro-1-hexanesulfonamide | $7.7\times10^{-2}$ | | Zhang et al. (2010) | Q | 287, 288 |
| $C_{11}H_{13}N_2O_2F_{13}S$ | 1.5 | | Zhang et al. (2010) | Q | 287, 289 |
| [50598-28-2] | $2.2\times10^{-1}$ | | Zhang et al. (2010) | Q | 287, 290 |
| INDOGKYZYLAGEM-UHFFFAOYSA-N | $2.5\times10^{1}$ | | Zhang et al. (2010) | Q | 287, 291 |



Table A9.1: Sulfur (C, H, O, N, Cl, S) (...continued)

| Substance Formula (Trivial Name) [CAS Registry Number] InChIKey | $H_s^{cp}$ (at $T^{\ominus}$) $\left[\dfrac{\text{mol}}{\text{m}^3\,\text{Pa}}\right]$ | $\dfrac{\text{d}\ln H_s^{cp}}{\text{d}(1/T)}$ [K] | Reference | Type | Note |
|---|---|---|---|---|---|
| 2-(methyl((undecafluoropentyl) sulfonyl)amino)ethyl prop-2-enoate | $9.7{\times}10^{-2}$ | | Zhang et al. (2010) | Q | 287, 288 |
| $C_{11}H_{10}F_{11}NO_4S$ | $2.0{\times}10^{-1}$ | | Zhang et al. (2010) | Q | 287, 289 |
| [67584-56-9] | $7.5{\times}10^{1}$ | | Zhang et al. (2010) | Q | 287, 290 |
| FZWFDJBZTLTRGH-UHFFFAOYSA-N | $9.7$ | | Zhang et al. (2010) | Q | 287, 291 |
| 2-(methyl((nonafluorobutyl) sulphonyl)amino)ethyl methacrylate | $3.3{\times}10^{-1}$ | | Zhang et al. (2010) | Q | 287, 288 |
| $C_{11}H_{12}F_9NO_4S$ | $5.0{\times}10^{-1}$ | | Zhang et al. (2010) | Q | 287, 289 |
| [67584-59-2] | $1.9{\times}10^{-1}$ | | Zhang et al. (2010) | Q | 287, 290 |
| BEIWUHUHJDEMQO-UHFFFAOYSA-N | $2.9{\times}10^{1}$ | | Zhang et al. (2010) | Q | 287, 291 |
| N-(3-(dimethylamino)propyl) pentadecafluoro-1-heptanesulfonamide | $1.5{\times}10^{-2}$ | | Zhang et al. (2010) | Q | 287, 288 |
| $C_{12}H_{13}F_{15}N_2O_2S$ | $2.3{\times}10^{-1}$ | | Zhang et al. (2010) | Q | 287, 289 |
| [67584-54-7] | $1.8{\times}10^{-2}$ | | Zhang et al. (2010) | Q | 287, 290 |
| RFJQYRXZGNILIY-UHFFFAOYSA-N | $5.2$ | | Zhang et al. (2010) | Q | 287, 291 |
| acrylic acid 2-[methyl[(tridecafluorohexyl)sulfonyl]amino]ethyl ester | $1.8{\times}10^{-2}$ | | Zhang et al. (2010) | Q | 287, 288 |
| $C_{12}H_{10}F_{13}NO_4S$ | $6.0{\times}10^{-2}$ | | Zhang et al. (2010) | Q | 287, 289 |
| [67584-57-0] | $3.4{\times}10^{1}$ | | Zhang et al. (2010) | Q | 287, 290 |
| HLKZFXXWGVPYAY-UHFFFAOYSA-N | $2.0$ | | Zhang et al. (2010) | Q | 287, 291 |
| N-ethyl perfluorooctane sulfonamidoethanol | $1.7{\times}10^{-2}$ | | Zhang et al. (2010) | Q | 287, 288 |
| $C_{12}H_{10}F_{17}NO_3S$ | $8.6{\times}10^{-3}$ | | Zhang et al. (2010) | Q | 287, 289 |
| (EtFOSE) | $6.2{\times}10^{-3}$ | | Zhang et al. (2010) | Q | 287, 290 |
| [1691-99-2] | $3.3{\times}10^{-1}$ | | Zhang et al. (2010) | Q | 287, 291 |
| HUFHNYZNTFSKCT-UHFFFAOYSA-N | $5.7{\times}10^{-2}$ | | Arp et al. (2006) | Q | 633 |
| | $1.2{\times}10^{-3}$ | | Arp et al. (2006) | Q | 634 |
| flumetsulam $C_{12}H_9F_2N_5O_2S$ [98967-40-9] RXCPQSJAVKGONC-UHFFFAOYSA-N | $1.5{\times}10^{11}$ | | Ebert et al. (2023) | ? | 316 |
| florasulam | $1.7{\times}10^{6}$ | | HSDB (2015) | V | |
| $C_{12}H_8F_3N_5O_3S$ | $2.3{\times}10^{6}$ | | Maniere et al. (2011) | ? | 12, 493, 165 |
| [145701-23-1] | $3.4{\times}10^{7}$ | | Maniere et al. (2011) | ? | 12, 573, 165 |
| QZXATCCPQKOEIH-UHFFFAOYSA-N | $3.0{\times}10^{4}$ | | Maniere et al. (2011) | ? | 12, 570, 165 |





Table A9.1: Sulfur (C, H, O, N, Cl, S) (... continued)

| Substance Formula (Trivial Name) [CAS Registry Number] InChIKey | $H_s^{cp}$ (at $T^{\ominus}$) $\left[\dfrac{\text{mol}}{\text{m}^3\,\text{Pa}}\right]$ | $\dfrac{\text{d}\ln H_s^{cp}}{\text{d}(1/T)}$ [K] | Reference | Type | Note |
|---|---|---|---|---|---|
| 2-(methyl((pentadecafluoroheptyl) sulphonyl)amino)ethyl acrylate | $3.5\times10^{-3}$ | | Zhang et al. (2010) | Q | 287, 288 |
| $C_{13}H_{10}F_{15}NO_4S$ | $1.5\times10^{-2}$ | | Zhang et al. (2010) | Q | 287, 289 |
| [68084-62-8] | 9.9 | | Zhang et al. (2010) | Q | 287, 290 |
| BEYGZFVOTUDDJK-UHFFFAOYSA-N | $4.2\times10^{-1}$ | | Zhang et al. (2010) | Q | 287, 291 |
| flazasulfuron | $1.6\times10^{6}$ | | HSDB (2015) | Q | 99 |
| $C_{13}H_{12}F_3N_5O_5S$ | $>3.9\times10^{5}$ | | Maniere et al. (2011) | ? | 165 |
| [104040-78-0] | | | | | |
| HWATZEJQIXKWQS-UHFFFAOYSA-N | | | | | |
| tritosulfuron | $>1.0\times10^{4}$ | | Maniere et al. (2011) | ? | 12, 165 |
| $C_{13}H_9F_6N_5O_4S$ | | | | | |
| [142469-14-5] | | | | | |
| KVEQCVKVIFQSGC-UHFFFAOYSA-N | | | | | |
| pyroxsulam | $1.4\times10^{6}$ | | Maniere et al. (2011) | ? | 241, 165 |
| $C_{14}H_{13}F_3N_6O_5S$ | | | | | |
| [422556-08-9] | | | | | |
| GLBLPMUBLHYFCW-UHFFFAOYSA-N | | | | | |
| N-methyl perfluorooctane sulfonamidoethylacrylate | $4.4\times10^{-2}$ | | Arp et al. (2006) | Q | 633 |
| $C_{14}H_{10}F_{17}NO_4S$ | $2.2\times10^{-3}$ | | Arp et al. (2006) | Q | 634 |
| (MeFOSEA) | | | | | |
| [25268-77-3] | | | | | |
| RTJZWOGSCLVJLD-UHFFFAOYSA-N | | | | | |
| pyrasulfotole | $7.0\times10^{8}$ | | HSDB (2015) | V | |
| $C_{14}H_{13}F_3N_2O_4S$ | | | | | |
| [365400-11-9] | | | | | |
| CZRVDACSCJKRFL-UHFFFAOYSA-N | | | | | |
| triafamone | $8.2\times10^{3}$ | | Ebert et al. (2023) | ? | 318 |
| $C_{14}H_{13}F_3N_4O_5S$ | | | | | |
| [874195-61-6] | | | | | |
| GBHVIWKSEHWFDD-UHFFFAOYSA-N | | | | | |
| flufenacet | $1.7\times10^{3}$ | | HSDB (2015) | V | |
| $C_{14}H_{13}F_4N_3O_2S$ | $1.1\times10^{3}$ | | Maniere et al. (2011) | ? | 12, 165 |
| [142459-58-3] | | | | | |
| IANUJLZYFUDJIH-UHFFFAOYSA-N | | | | | |





Table A9.1: Sulfur (C, H, O, N, Cl, S) (...continued)

| Substance<br>Formula<br>(Trivial Name)<br>[CAS Registry Number]<br>InChIKey | $H_s^{cp}$ (at $T^{\ominus}$) $\left[\dfrac{\mathrm{mol}}{\mathrm{m}^3\,\mathrm{Pa}}\right]$ | $\dfrac{\mathrm{d}\ln H_s^{cp}}{\mathrm{d}(1/T)}$ [K] | Reference | Type | Note |
|---|---|---|---|---|---|
| N-butyl-1,1,2,2,3,3,4,4,5,5,6,6,7,7,8,8,8-heptadecafluoro-N-(2-hydroxyethyl)-1-octanesulfonamide | $9.7{\times}10^{-3}$ | | Zhang et al. (2010) | Q | 287, 288 |
| $C_{14}H_{14}F_{17}NO_3S$ | $4.1{\times}10^{-3}$ | | Zhang et al. (2010) | Q | 287, 289 |
| [2263-09-4] | $3.8{\times}10^{-2}$ | | Zhang et al. (2010) | Q | 287, 290 |
| AQWROZBAYZBWIH-UHFFFAOYSA-N | $2.2{\times}10^{-1}$ | | Zhang et al. (2010) | Q | 287, 291 |
| ethyl N-ethyl-N-[(heptadecafluorooctyl)sulphonyl]glycinate | $1.3{\times}10^{-5}$ | | Zhang et al. (2010) | Q | 287, 288 |
| $C_{14}H_{12}NO_4F_{17}S$ | $1.8{\times}10^{-3}$ | | Zhang et al. (2010) | Q | 287, 289 |
| [1869-77-8] | 4.3 | | Zhang et al. (2010) | Q | 287, 290 |
| LMUUXHHNCDERBQ-UHFFFAOYSA-N | $3.0{\times}10^{-2}$ | | Zhang et al. (2010) | Q | 287, 291 |
| perfluidone | $1.3{\times}10^{2}$ | | Duchowicz et al. (2020) | V | 186 |
| $C_{14}H_{12}F_3NO_4S_2$ | $1.6{\times}10^{8}$ | | Duchowicz et al. (2020) | Q | |
| [37924-13-3] | | | | | |
| WHTBVLXUSXVMEV-UHFFFAOYSA-N | | | | | |
| 2-(((heptadecafluorooctyl)sulfonyl)methylamino)ethyl methacrylate | $4.2{\times}10^{-4}$ | | Zhang et al. (2010) | Q | 287, 288 |
| $C_{15}H_{12}F_{17}NO_4S$ | $3.3{\times}10^{-3}$ | | Zhang et al. (2010) | Q | 287, 289 |
| [14650-24-9] | 2.5 | | Zhang et al. (2010) | Q | 287, 290 |
| UZMOXNBUTMPDCX-UHFFFAOYSA-N | $5.4{\times}10^{-2}$ | | Zhang et al. (2010) | Q | 287, 291 |
| 2-(N-ethylperfluorooctane-sulfonamido)ethyl acrylate | 1.1 | | Ebert et al. (2023) | ? | 371 |
| $C_{15}H_{12}F_{17}NO_4S$ | | | | | |
| [423-82-5] | | | | | |
| ZAZJGBCGMUKZEL-UHFFFAOYSA-N | | | | | |
| isoxaflutole | $5.3{\times}10^{4}$ | | MacBean (2012b) | X | 350 |
| $C_{15}H_{12}F_3NO_4S$ | $5.3{\times}10^{4}$ | | Maniere et al. (2011) | ? | 12, 165 |
| [141112-29-0] | | | | | |
| OYIKARCXOQLFHF-UHFFFAOYSA-N | | | | | |
| primisulfuron-methyl | $7.0{\times}10^{6}$ | | HSDB (2015) | Q | 99 |
| $C_{15}H_{12}F_4N_4O_7S$ | | | | | |
| [86209-51-0] | | | | | |
| ZTYVMAQSHCZXLF-UHFFFAOYSA-N | | | | | |
| dithiopyr | 6.5 | | Ebert et al. (2023) | ? | 739 |
| $C_{15}H_{16}F_5NO_2S_2$ | | | | | |
| [97886-45-8] | | | | | |
| YUBJPYNSGLJZPQ-UHFFFAOYSA-N | | | | | |



Table A9.1: Sulfur (C, H, O, N, Cl, S) (. . . continued)

| Substance Formula (Trivial Name) [CAS Registry Number] InChIKey | $H_s^{cp}$ (at $T^{\ominus}$) $\left[\dfrac{\text{mol}}{\text{m}^3\,\text{Pa}}\right]$ | $\dfrac{\text{d}\ln H_s^{cp}}{\text{d}(1/T)}$ [K] | Reference | Type | Note |
|---|---|---|---|---|---|
| penoxsulam | $9.0\times10^{12}$ | | HSDB (2015) | V | |
| $C_{16}H_{14}F_5N_5O_5S$ | $1.2\times10^{14}$ | | Maniere et al. (2011) | ? | 241, 573, 165 |
| [219714-96-2] | $4.7\times10^{11}$ | | Maniere et al. (2011) | ? | 241, 570, 165 |
| SYJGKVOENHZYMQ-UHFFFAOYSA-N | $3.4\times10^{13}$ | | Maniere et al. (2011) | ? | 241, 493, 165 |
| pantoprazole $C_{16}H_{15}F_2N_3O_4S$ [102625-70-7] IQPSEEYGBUAQFF-UHFFFAOYSA-N | $1.7\times10^{14}$ | | HSDB (2015) | Q | 99 |
| 2-(N-ethylperfluorooctanesulfamido)ethyl methacrylate | $3.2\times10^{-4}$ | | Zhang et al. (2010) | Q | 287, 288 |
| $C_{16}H_{14}F_{17}NO_4S$ | $3.9\times10^{-3}$ | | Zhang et al. (2010) | Q | 287, 289 |
| [376-14-7] | $1.5$ | | Zhang et al. (2010) | Q | 287, 290 |
| DBCGADAHIXJHCE-UHFFFAOYSA-N | $4.1\times10^{-2}$ | | Zhang et al. (2010) | Q | 287, 291 |
| thiazopyr | $2.1\times10^{1}$ | | Duchowicz et al. (2020) | V | 186 |
| $C_{16}H_{17}F_5N_2O_2S$ | $2.1\times10^{1}$ | | HSDB (2015) | V | |
| [117718-60-2] | $8.7\times10^{3}$ | | Duchowicz et al. (2020) | Q | |
| YIJZJEYQBAAWRJ-UHFFFAOYSA-N | | | | | |
| penthiopyrad | $7.1\times10^{3}$ | | Maniere et al. (2011) | ? | 12, 570, 165 |
| $C_{16}H_{20}F_3N_3OS$ | $1.3\times10^{2}$ | | Maniere et al. (2011) | ? | 12, 493, 165 |
| [183675-82-3] | $2.4\times10^{3}$ | | Maniere et al. (2011) | ? | 12, 577, 165 |
| PFFIDZXUXFLSSR-UHFFFAOYSA-N | $1.6\times10^{2}$ | | Maniere et al. (2011) | ? | 12, 572, 165 |
| triflusulfuron-methyl $C_{17}H_{19}F_3N_6O_6S$ [126535-15-7] IMEVJVISCHQJRM-UHFFFAOYSA-N | $>4.2\times10^{2}$ | | Maniere et al. (2011) | ? | 241, 577, 165 |
| celecoxib $C_{17}H_{14}F_3N_3O_2S$ [169590-42-5] RZEKVGVHFLEQIL-UHFFFAOYSA-N | $1.3\times10^{7}$ | | HSDB (2015) | Q | 99 |



Table A9.1: Sulfur (C, H, O, N, Cl, S) (...continued)

| Substance<br>Formula<br>(Trivial Name)<br>[CAS Registry Number]<br>InChIKey | $H_s^{cp}$<br>(at $T^{\ominus}$)<br>$\left[\dfrac{\mathrm{mol}}{\mathrm{m}^3\,\mathrm{Pa}}\right]$ | $\dfrac{\mathrm{d}\ln H_s^{cp}}{\mathrm{d}(1/T)}$<br><br>[K] | Reference | Type | Note |
|---|---|---|---|---|---|
| 2-(butyl((heptadecafluorooctyl)sulfonyl)amino)ethyl acrylate | $2.9\times10^{-4}$ | | Zhang et al. (2010) | Q | 287, 288 |
| $C_{17}H_{16}F_{17}NO_4S$ | $4.1\times10^{-3}$ | | Zhang et al. (2010) | Q | 287, 289 |
| [383-07-3] | 1.8 | | Zhang et al. (2010) | Q | 287, 290 |
| AQQNNAXBGPWALO-UHFFFAOYSA-N | $4.3\times10^{-2}$ | | Zhang et al. (2010) | Q | 287, 291 |
| thidiazimin | $3.5\times10^{8}$ | | MacBean (2012a) | ? | |
| $C_{18}H_{17}N_4O_2FS$ | | | | | |
| [123249-43-4] | | | | | |
| HZKBYBNLTLVSPX-JZJYNLBNSA-N | | | | | |
| benthiavalicarb isopropyl | $1.1\times10^{2}$ | | MacBean (2012b) | X | 350 |
| $C_{18}H_{24}FN_3O_3S$ | $>1.1\times10^{2}$ | | Maniere et al. (2011) | ? | 12, 165 |
| [177406-68-7] | | | | | |
| USRKFGIXLGKMKU-IAQYHMDHSA-N | | | | | |
| flutianil | $7.2\times10^{1}$ | | Ebert et al. (2023) | ? | 318 |
| $C_{19}H_{14}F_4N_2OS_2$ | | | | | |
| [958647-10-4] | | | | | |
| KGXUEPOHGFWQKF-ZCXUNETKSA-N | | | | | |
| rosuvastatin | $2.9\times10^{14}$ | | HSDB (2015) | Q | 99 |
| $C_{22}H_{28}N_3O_6FS$ | | | | | |
| [287714-41-4] | | | | | |
| BPRHUIZQVSMCRT-YXWZHEERSA-N | | | | | |
| oxathiapiprolin | $2.8\times10^{2}$ | | Maniere et al. (2011) | ? | 12, 165 |
| $C_{24}H_{22}F_5N_5O_2S$ | | | | | |
| [1003318-67-9] | | | | | |
| IAQLCKZJGNTRDO-UHFFFAOYSA-N | | | | | |
| trichloromethanesulfenyl chloride | $4.1\times10^{-2}$ | | Zhang et al. (2010) | Q | 287, 288 |
| $CCl_4S$ | $6.9\times10^{-4}$ | | Zhang et al. (2010) | Q | 287, 289 |
| [594-42-3] | $9.5\times10^{-4}$ | | Zhang et al. (2010) | Q | 287, 290 |
| RYFZYYUIAZYQLC-UHFFFAOYSA-N | $5.3\times10^{-3}$ | | Zhang et al. (2010) | Q | 287, 291 |
| 1,1,2,2-tetrachloroethanesulfenyl chloride | $8.8\times10^{-2}$ | | Zhang et al. (2010) | Q | 287, 288 |
| $C_2HCl_5S$ | $3.7\times10^{-3}$ | | Zhang et al. (2010) | Q | 287, 289 |
| [1185-09-7] | $1.9\times10^{-2}$ | | Zhang et al. (2010) | Q | 287, 290 |
| LCVOCDOSGJHZFH-UHFFFAOYSA-N | $6.7\times10^{-2}$ | | Zhang et al. (2010) | Q | 287, 291 |
| 2-chloroethyl methyl sulfide | $5.3\times10^{-2}$ | | Bartelt-Hunt et al. (2008) | ? | 21 |
| $C_3H_7ClS$ | | | | | |
| [542-81-4] | | | | | |
| MYFKLQFBFSHBPA-UHFFFAOYSA-N | | | | | |



Table A9.1: Sulfur (C, H, O, N, Cl, S) (. . . continued)

| Substance Formula (Trivial Name) [CAS Registry Number] InChIKey | $H_s^{cp}$ (at $T^\ominus$) $\left[\dfrac{\text{mol}}{\text{m}^3\,\text{Pa}}\right]$ | $\dfrac{\text{d}\ln H_s^{cp}}{\text{d}(1/T)}$ [K] | Reference | Type | Note |
|---|---|---|---|---|---|
| 2-chloroethyl ethyl sulfide<br>$C_4H_9ClS$<br>[693-07-2]<br>GBNVXYXIRHSYEG-UHFFFAOYSA-N | $2.0\times10^{-2}$<br>$2.7\times10^{-2}$ | | HSDB (2015)<br>Bartelt-Hunt et al. (2008) | Q<br>? | 99<br>21 |
| 2,2'-dichlorodiethylsulfide<br>$(ClCH_2CH_2)_2S$<br>(mustard gas)<br>[505-60-2]<br>QKSKPIVNLNLAAV-UHFFFAOYSA-N | $3.0\times10^{-1}$<br>$2.2\times10^{-2}$<br>$1.5\times10^{-1}$<br>$1.6\times10^{-1}$<br>$4.0\times10^{-1}$<br>$4.1\times10^{-1}$<br>$4.1\times10^{-1}$ | | Hine and Mookerjee (1975)<br>Keshavarz et al. (2022)<br>Duchowicz et al. (2020)<br>Modarresi et al. (2007)<br>Duchowicz et al. (2020)<br>Bartelt-Hunt et al. (2008)<br>Opresko et al. (1998) | V<br>Q<br>Q<br>Q<br>?<br>?<br>? | <br><br>299<br>67<br>185, 21<br>21<br> |
| 1,2-bis(2-chloroethylthio)ethane<br>$C_6H_{12}Cl_2S_2$<br>(sesquimustard)<br>[3563-36-8]<br>AMGNHZVUZWILSB-UHFFFAOYSA-N | $8.8\times10^1$ | | Ebert et al. (2023) | ? | 318 |
| pentachlorobenzenethiol<br>$C_6HCl_5S$<br>[133-49-3]<br>LLMLGZUZTFMXSA-UHFFFAOYSA-N | $6.6\times10^{-2}$<br>$8.4\times10^{-2}$<br>$2.7\times10^{-2}$<br>$1.3$<br>$2.2\times10^{-2}$ | | HSDB (2015)<br>Zhang et al. (2010)<br>Zhang et al. (2010)<br>Zhang et al. (2010)<br>Zhang et al. (2010) | Q<br>Q<br>Q<br>Q<br>Q | 99<br>287, 288<br>287, 289<br>287, 290<br>287, 291 |
| chloroethyl phenyl sulfide<br>$C_8H_9ClS$<br>[5535-49-9]<br>QDXIHHOPZFCEAP-UHFFFAOYSA-N | $1.3\times10^{-1}$ | | Bartelt-Hunt et al. (2008) | ? | 21 |
| tetrasul<br>$C_{12}H_6Cl_4S$<br>[2227-13-6]<br>QUWSDLYBOVGOCW-UHFFFAOYSA-N | $9.3\times10^{-1}$ | | MacBean (2012a) | ? | |
| methanesulfonyl chloride<br>$CH_3ClO_2S$<br>[124-63-0]<br>QARBMVPHQWIHKH-UHFFFAOYSA-N | $2.2\times10^{-1}$ | | HSDB (2015) | Q | 99 |
| bis(trichloromethyl)sulfone<br>$C_2Cl_6O_2S$<br>[3064-70-8]<br>YBNLWIZAWPBUKQ-UHFFFAOYSA-N | $8.2\times10^2$<br>$8.2\times10^2$<br>$1.2\times10^{-2}$<br>$3.1\times10^3$<br>$1.0\times10^1$ | | HSDB (2015)<br>Zhang et al. (2010)<br>Zhang et al. (2010)<br>Zhang et al. (2010)<br>Zhang et al. (2010) | Q<br>Q<br>Q<br>Q<br>Q | 99<br>287, 288<br>287, 289<br>287, 290<br>287, 291 |
| benzenesulfonyl chloride<br>$C_6H_5ClO_2S$<br>[98-09-9]<br>CSKNSYBAZOQPLR-UHFFFAOYSA-N | $1.1$<br>$1.7\times10^1$<br>$6.7$<br>$1.6\times10^2$ | | Zhang et al. (2010)<br>Zhang et al. (2010)<br>Zhang et al. (2010)<br>Zhang et al. (2010) | Q<br>Q<br>Q<br>Q | 287, 288<br>287, 289<br>287, 290<br>287, 291 |



Table A9.1: Sulfur (C, H, O, N, Cl, S) (...continued)

| Substance Formula (Trivial Name) [CAS Registry Number] InChIKey | $H_s^{cp}$ (at $T^\ominus$) $\left[\dfrac{\text{mol}}{\text{m}^3\,\text{Pa}}\right]$ | $\dfrac{\text{d}\ln H_s^{cp}}{\text{d}(1/T)}$ [K] | Reference | Type | Note |
|---|---|---|---|---|---|
| 4-methylbenzenesulfonyl chloride | 1.0 | | Zhang et al. (2010) | Q | 287, 288 |
| C$_7$H$_7$ClO$_2$S | $1.8\times10^1$ | | Zhang et al. (2010) | Q | 287, 289 |
| [98-59-9] | $1.2\times10^1$ | | Zhang et al. (2010) | Q | 287, 290 |
| YYROPELSRYBVMQ-UHFFFAOYSA-N | $9.2\times10^1$ | | Zhang et al. (2010) | Q | 287, 291 |
| endosulfan | $9.4\times10^{-1}$ | | Mackay et al. (2006d) | V | |
| C$_9$H$_6$Cl$_6$O$_3$S | $1.1\times10^2$ | | Zhang et al. (2010) | Q | 287, 288 |
| [115-29-7] | $2.4\times10^1$ | | Zhang et al. (2010) | Q | 287, 289 |
| RDYMFSUJUZBWLH-UHFFFAOYSA-N | $1.1\times10^2$ | | Zhang et al. (2010) | Q | 287, 290 |
| | $2.3\times10^8$ | | Zhang et al. (2010) | Q | 287, 291 |
| | $3.1\times10^1$ | | Hilal et al. (2008) | Q | |
| | $7.8\times10^{-1}$ | | Modarresi et al. (2007) | Q | 67 |
| $\alpha$-endosulfan | 1.4 | | Shen and Wania (2005) | L | 366 |
| C$_9$H$_6$Cl$_6$O$_3$S | 1.4 | | Shen and Wania (2005) | L | 367 |
| (endosulfan I) | 1.4 | | Muir et al. (2004) | L | 367 |
| [959-98-8] | 1.4 | | Muir et al. (2004) | L | 366 |
| RDYMFSUJUZBWLH-AMHWMVONSA-N | 1.2 | | Chao et al. (2017) | M | |
| | 1.3 | 4200 | Cetin et al. (2006) | M | |
| | 1.4 | | Altschuh et al. (1999) | M | |
| | $1.5\times10^{-1}$ | | Rice et al. (1997b) | M | 12 |
| | $1.3\times10^{-1}$ | 2300 | Rice et al. (1997a) | M | |
| | 1.5 | | Cotham and Bidleman (1989) | V | |
| | $3.4\times10^{-1}$ | | Suntio et al. (1988) | V | 12 |
| | $9.2\times10^{-1}$ | | Suntio et al. (1988) | C | |
| | $9.6\times10^{-2}$ | | Keshavarz et al. (2022) | Q | |
| | $5.2\times10^1$ | | Duchowicz et al. (2020) | Q | |
| | $8.0\times10^{-1}$ | | Modarresi et al. (2007) | Q | 67 |
| | | 3200 | Kühne et al. (2005) | Q | |
| | 1.4 | | Duchowicz et al. (2020) | ? | 185, 21 |
| | | 2300 | Kühne et al. (2005) | ? | |
| $\beta$-endosulfan | $2.5\times10^1$ | | Shen and Wania (2005) | L | 366 |
| C$_9$H$_6$Cl$_6$O$_3$S | $2.2\times10^1$ | | Shen and Wania (2005) | L | 367 |
| (endosulfan II) | 6.4 | | Chao et al. (2017) | M | |
| [33213-65-9] | $1.9\times10^1$ | 3700 | Cetin et al. (2006) | M | |
| RDYMFSUJUZBWLH-MDBBVBRHSA-N | 2.5 | | Altschuh et al. (1999) | M | |
| | 1.1 | | Rice et al. (1997b) | M | 12 |
| | 1.1 | | Rice et al. (1997a) | M | 12 |
| | $1.6\times10^1$ | | Cotham and Bidleman (1989) | V | |
| | $1.6\times10^3$ | | Keshavarz et al. (2022) | Q | |
| | $5.2\times10^1$ | | Duchowicz et al. (2020) | Q | 184 |
| | $3.1\times10^1$ | | Hilal et al. (2008) | Q | |
| | $8.0\times10^{-1}$ | | Modarresi et al. (2007) | Q | 67 |
| | $2.5\times10^1$ | | Duchowicz et al. (2020) | ? | 185, 21 |



Table A9.1: Sulfur (C, H, O, N, Cl, S) (... continued)

| Substance Formula (Trivial Name) [CAS Registry Number] InChIKey | $H_s^{cp}$ (at $T^{\ominus}$) $\left[\dfrac{\text{mol}}{\text{m}^3\,\text{Pa}}\right]$ | $\dfrac{\text{d}\ln H_s^{cp}}{\text{d}(1/T)}$ [K] | Reference | Type | Note |
|---|---|---|---|---|---|
| endosulfan sulfate $C_9H_6Cl_6O_4S$ [1031-07-8] AAPVQEMYVNZIOO-UHFFFAOYSA-N | $8.2\times10^5$ | | HSDB (2015) | V | |
| mcpa-thioethyl $C_{11}H_{13}ClO_2S$ [25319-90-8] AZFKQCNGMSSWDS-UHFFFAOYSA-N | $4.5\times10^{-1}$ | | Mackay et al. (2006d) | V | |
| 1,1'-sulfonylbis(4-chlorobenzene) $C_{12}H_8Cl_2O_2S$ [80-07-9] GPAPPPVRLPGFEQ-UHFFFAOYSA-N | $7.0\times10^1$ $7.2\times10^1$ $6.5\times10^3$ $5.0\times10^4$ $3.1\times10^3$ | | HSDB (2015) Zhang et al. (2010) Zhang et al. (2010) Zhang et al. (2010) Zhang et al. (2010) | Q Q Q Q Q | 99 287, 288 287, 289 287, 290 287, 291 |
| 2,2'-thiobis-4,6-dichlorophenol $C_{12}H_6Cl_4O_2S$ (bithionol) [97-18-7] JFIOVJDNOJYLKP-UHFFFAOYSA-N | $1.1\times10^5$ | | Ebert et al. (2023) | ? | 318 |
| 1,2,4-trichloro-5-[(4-chlorophenyl)sulfonyl]benzene $C_{12}H_6Cl_4O_2S$ (tetradifon) [116-29-0] MLGCXEBRWGEOQX-UHFFFAOYSA-N | $6.9\times10^3$ $6.9\times10^3$ $1.0\times10^3$ | | Duchowicz et al. (2020) HSDB (2015) Duchowicz et al. (2020) | V V Q | 186 |
| ovex $C_{12}H_8Cl_2O_3S$ (chlorfenson) [80-33-1] RZXLPPRPEOUENN-UHFFFAOYSA-N | $6.2\times10^1$ | | HSDB (2015) | Q | 99 |
| sulphenone $C_{12}H_9ClO_2S$ [80-00-2] OFCFYWOKHPOXKF-UHFFFAOYSA-N | $5.2\times10^1$ | | HSDB (2015) | Q | 99 |
| sulcotrione $C_{14}H_{13}ClO_5S$ [99105-77-8] PQTBTIFWAXVEPB-UHFFFAOYSA-N | $1.7\times10^6$ | | Maniere et al. (2011) | ? | 12, 816, 165 |
| aramite $C_{15}H_{23}ClO_4S$ [140-57-8] YKFRAOGHWKADFJ-UHFFFAOYSA-N | $5.2\times10^1$ | | HSDB (2015) | Q | 99 |



Table A9.1: Sulfur (C, H, O, N, Cl, S) (... continued)

| Substance<br>Formula<br>(Trivial Name)<br>[CAS Registry Number]<br>InChIKey | $H_s^{cp}$<br>(at $T^{\ominus}$)<br>$\left[\dfrac{\mathrm{mol}}{\mathrm{m^3\,Pa}}\right]$ | $\dfrac{\mathrm{d}\ln H_s^{cp}}{\mathrm{d}(1/T)}$<br><br>[K] | Reference | Type | Note |
|---|---|---|---|---|---|
| 6-chloro-2-(6-chloro-4-methyl-3-<br>oxobenzo[*b*]thien-2(3H)-ylidene)-<br>4-methylbenzo[*b*]thiophene-3(2H)-<br>one | $3.2\times10^7$ | | Zhang et al. (2010) | Q | 287, 288 |
| $C_{18}H_{10}Cl_2O_2S_2$ | $1.2\times10^7$ | | Zhang et al. (2010) | Q | 287, 289 |
| [2379-74-0] | $5.4\times10^4$ | | Zhang et al. (2010) | Q | 287, 290 |
| NDDLLTAIKYHPOD-ISLYRVAYSA-N | $2.4\times10^6$ | | Zhang et al. (2010) | Q | 287, 291 |
| 5-chloro-3-(trichloromethyl)-1,2,4-<br>thiadiazole | $1.6\times10^1$ | | Zhang et al. (2010) | Q | 287, 288 |
| $C_3Cl_4N_2S$ | $1.9\times10^1$ | | Zhang et al. (2010) | Q | 287, 289 |
| [5848-93-1] | $7.3\times10^{-1}$ | | Zhang et al. (2010) | Q | 287, 290 |
| MARKPJMFLDWCID-UHFFFAOYSA-N | $4.2\times10^{-1}$ | | Zhang et al. (2010) | Q | 287, 291 |
| 2,4-dichloro-6-(methylthio)-1,3,5-<br>triazine | $1.3\times10^1$ | | Zhang et al. (2010) | Q | 287, 288 |
| $C_4H_3Cl_2N_3S$ | $1.3\times10^1$ | | Zhang et al. (2010) | Q | 287, 289 |
| [13705-05-0] | $9.7\times10^{-1}$ | | Zhang et al. (2010) | Q | 287, 290 |
| MWPZLWRHHPWTFS-UHFFFAOYSA-N | $1.7\times10^1$ | | Zhang et al. (2010) | Q | 287, 291 |
| chlorthiamid<br>$C_7H_5Cl_2NS$<br>[1918-13-4]<br>KGKGSIUWJCAFPX-UHFFFAOYSA-N | $3.5\times10^4$ | | MacBean (2012a) | ? | |
| (2-chlorophenyl)thiourea<br>$C_7H_7ClN_2S$<br>[5344-82-1]<br>YZUKKTCDYSIWKJ-UHFFFAOYSA-N | $>9.9\times10^1$ | | HSDB (2015) | Q | 545 |
| 2-chloroallyl-N,N-<br>diethyldithiocarbamate | $1.5$ | | Duchowicz et al. (2020) | V | 186 |
| $C_8H_{14}ClNS_2$ | $1.5$ | | HSDB (2015) | V | |
| [95-06-7] | $2.1\times10^2$ | | Duchowicz et al. (2020) | Q | |
| XJCLWVXTCRQIDI-UHFFFAOYSA-N | $2.1\times10^1$ | | Hilal et al. (2008) | Q | |
| | $4.1$ | | Modarresi et al. (2007) | Q | 67 |
| thiacloprid<br>$C_{10}H_9ClN_4S$ | $9.0\times10^8$ | | HSDB (2015) | V | |
| [111988-49-9] | $2.1\times10^9$ | | Maniere et al. (2011) | ? | 241, 165 |
| HOKKPVIRMVDYPB-UHFFFAOYSA-N | | | | | |
| chloromethiuron<br>$C_{10}H_{13}N_2ClS$<br>[28217-97-2]<br>IBZZDPVVVSNQOY-UHFFFAOYSA-N | $2.0\times10^5$ | | MacBean (2012a) | ? | |
| imibenconazole | $4.9\times10^4$ | | Duchowicz et al. (2020) | V | 186 |
| $C_{17}H_{13}Cl_3N_4S$ | $6.8\times10^3$ | | Duchowicz et al. (2020) | Q | |
| [86598-92-7]<br>AGKSTYPVMZODRV-UHFFFAOYSA-N | | | | | |



Table A9.1: Sulfur (C, H, O, N, Cl, S) (. . . continued)

| Substance Formula (Trivial Name) [CAS Registry Number] InChIKey | $H_s^{cp}$ (at $T^\ominus$) $\left[\dfrac{\text{mol}}{\text{m}^3\,\text{Pa}}\right]$ | $\dfrac{\text{d}\ln H_s^{cp}}{\text{d}(1/T)}$ [K] | Reference | Type | Note |
|---|---|---|---|---|---|
| etridiazole $C_5H_5Cl_3N_2OS$ [2593-15-9] KQTVWCSONPJJPE-UHFFFAOYSA-N | $3.6\times10^1$ $3.3\times10^{-1}$ $1.6\times10^1$ $5.3\times10^1$ | | Duchowicz et al. (2020) HSDB (2015) Mackay et al. (2006d) Duchowicz et al. (2020) | V V V Q | 186 |
| 4-chloro-3-nitrobenzenesulfonamide $C_6H_5ClN_2O_4S$ [97-09-6] SPZGXONNVLTQDE-UHFFFAOYSA-N | $8.2\times10^3$ | | HSDB (2015) | Q | 99 |
| clothianidin $C_6H_8ClN_5O_2S$ [210880-92-5] PGOOBECODWQEAB-UHFFFAOYSA-N | $3.4\times10^{10}$ $3.4\times10^{10}$ | | MacBean (2012b) Maniere et al. (2011) | X ? | 350 12, 165 |
| chlobenthiazone $C_8H_6NOClS$ [63755-05-5] QCPASDYEQAVIJF-UHFFFAOYSA-N | $1.3$ | | MacBean (2012a) | ? | |
| prothiocarb hydrochloride $C_8H_{19}ClN_2OS$ [19622-19-6] NMFAMPYSJHIYMR-UHFFFAOYSA-N | $2.5\times10^9$ | | MacBean (2012a) | ? | |
| thicyofen $C_8H_5N_2OClS_2$ [116170-30-0] GNOOAFGERMHQJE-UHFFFAOYSA-N | $>2.3\times10^{10}$ | | MacBean (2012a) | ? | |
| thiamethoxam $C_8H_{10}ClN_5O_3S$ [153719-23-4] NWWZPOKUUAIXIW-DHZHZOJOSA-N | $2.1\times10^9$ $2.1\times10^9$ | | HSDB (2015) Maniere et al. (2011) | V ? | 165 |
| 4-amino-3,5-dichloro-N-ethyl-2-methylbenzenesulfonamide $C_9H_{12}Cl_2N_2O_2S$ [151574-12-8] SBBHTEMLRJJIGK-UHFFFAOYSA-N | $3.8\times10^4$ $1.1\times10^4$ $1.2\times10^6$ $1.5\times10^7$ | | Zhang et al. (2010) Zhang et al. (2010) Zhang et al. (2010) Zhang et al. (2010) | Q Q Q Q | 287, 288 287, 289 287, 290 287, 291 |
| dichlofluanid $C_9H_{11}Cl_2FN_2O_2S_2$ [1085-98-9] WURGXGVFSMYFCG-UHFFFAOYSA-N | $2.6\times10^2$ $1.9\times10^2$ $2.5\times10^4$ $5.7\times10^2$ $1.5\times10^1$ | | Duchowicz et al. (2020) Mackay et al. (2006d) Siebers and Mattusch (1996) Duchowicz et al. (2020) HSDB (2015) | V V V Q Q | 186 12 99 |



Table A9.1: Sulfur (C, H, O, N, Cl, S) (... continued)

| Substance / Formula / (Trivial Name) / [CAS Registry Number] / InChIKey | $H_s^{cp}$ (at $T^{\ominus}$) $\left[\dfrac{\mathrm{mol}}{\mathrm{m}^3\,\mathrm{Pa}}\right]$ | $\dfrac{\mathrm{d}\ln H_s^{cp}}{\mathrm{d}(1/T)}$ [K] | Reference | Type | Note |
|---|---|---|---|---|---|
| captan | $1.4\times10^3$ | | HSDB (2015) | V | |
| $C_9H_8Cl_3NO_2S$ | $1.5\times10^3$ | | Mackay et al. (2006d) | V | |
| [133-06-2] | 1.7 | | Suntio et al. (1988) | V | 12 |
| LDVVMCZRFWMZSG-UHFFFAOYSA-N | $1.6\times10^{-2}$ | | Barcelo and Hennion (1997) | X | 567 |
| | $2.1\times10^{-3}$ | | Goodarzi et al. (2010) | Q | 568, 571 |
| | $3.3\times10^3$ | | Maniere et al. (2011) | ? | 12, 570, 165 |
| | $5.0\times10^3$ | | Maniere et al. (2011) | ? | 12, 493, 165 |
| benazolin | $4.2\times10^6$ | | Ebert et al. (2023) | ? | 316 |
| $C_9H_6ClNO_3S$ | | | | | |
| [3813-05-6] | | | | | |
| HYJSGOXICXYZGS-UHFFFAOYSA-N | | | | | |
| folpet | $1.3\times10^2$ | | Duchowicz et al. (2020) | V | 186 |
| $C_9H_4Cl_3NO_2S$ | $1.3\times10^2$ | | HSDB (2015) | V | |
| [133-07-3] | 2.6 | | Mackay et al. (2006d) | V | |
| HKIOYBQGHSTUDB-UHFFFAOYSA-N | $2.6\times10^{-2}$ | | Barcelo and Hennion (1997) | X | 567 |
| | $1.9\times10^1$ | | Duchowicz et al. (2020) | Q | |
| | $1.2\times10^{-1}$ | | Goodarzi et al. (2010) | Q | 568 |
| | $1.2\times10^2$ | | Maniere et al. (2011) | ? | 165 |
| captafol | $2.0\times10^3$ | | Duchowicz et al. (2020) | V | 186 |
| $C_{10}H_9Cl_4NO_2S$ | $3.7\times10^3$ | | HSDB (2015) | V | |
| (difolatan) | $3.7\times10^1$ | | Duchowicz et al. (2020) | Q | |
| [2425-06-1] | | | | | |
| JHRWWRDRBPCWTF-UHFFFAOYSA-N | | | | | |
| diallate | 2.6 | | HSDB (2015) | V | |
| $C_{10}H_{17}Cl_2NOS$ | 9.3 | | Mackay et al. (2006d) | V | |
| (avadex) | 4.0 | | Suntio et al. (1988) | V | 12 |
| [2303-16-4] | 2.6 | | MacBean (2012a) | ? | |
| SPANOECCGNXGNR-UITAMQMPSA-N | | | | | |
| triallate | $8.2\times10^{-1}$ | | HSDB (2015) | V | |
| $C_{10}H_{16}Cl_3NOS$ | $8.8\times10^{-1}$ | | Mackay et al. (2006d) | V | |
| [2303-17-5] | $9.8\times10^{-1}$ | | Suntio et al. (1988) | V | 12 |
| MWBPRDONLNQCFV-UHFFFAOYSA-N | $1.0\times10^{-2}$ | | Barcelo and Hennion (1997) | X | 567 |
| | $1.6\times10^{-2}$ | | Goodarzi et al. (2010) | Q | 568 |
| | 1.1 | | Maniere et al. (2011) | ? | 12, 165 |
| tolylfluanid | $1.3\times10^1$ | | Duchowicz et al. (2020) | V | 186 |
| $C_{10}H_{13}Cl_2FN_2O_2S_2$ | $1.3\times10^1$ | | HSDB (2015) | V | |
| [731-27-1] | $1.6\times10^2$ | | Mackay et al. (2006d) | V | |
| HYVWIQDYBVKITD-UHFFFAOYSA-N | $3.7\times10^2$ | | Duchowicz et al. (2020) | Q | |



Table A9.1: Sulfur (C, H, O, N, Cl, S) (... continued)

| Substance<br>Formula<br>(Trivial Name)<br>[CAS Registry Number]<br>InChIKey | $H_s^{cp}$<br>(at $T^{\ominus}$)<br>$\left[\dfrac{\text{mol}}{\text{m}^3\,\text{Pa}}\right]$ | $\dfrac{\mathrm{d}\ln H_s^{cp}}{\mathrm{d}(1/T)}$<br><br>[K] | Reference | Type | Note |
|---|---|---|---|---|---|
| tiadinil<br>$C_{11}H_{10}ClN_3OS$<br>[223580-51-6]<br>VJQYLJSMBWXGDV-UHFFFAOYSA-N | $5.3\times10^4$ | | Ebert et al. (2023) | ? | 318 |
| benazolin-ethyl<br>$C_{11}H_{10}NO_3ClS$<br>[25059-80-7]<br>WQRCEBAZAUAUQC-UHFFFAOYSA-N | $4.7\times10^2$<br>$5.2\times10^3$ | | Duchowicz et al. (2020)<br>Duchowicz et al. (2020) | V<br>Q | 186 |
| orbencarb<br>$C_{12}H_{16}ClNOS$<br>[34622-58-7]<br>LLLFASISUZUJEQ-UHFFFAOYSA-N | $2.0\times10^1$ | | Ebert et al. (2023) | ? | 318 |
| S-(4-chlorobenzyl)<br>diethylthiocarbamate<br>$C_{12}H_{16}ClNOS$<br>(thiobencarb)<br>[28249-77-6]<br>QHTQREMOGMZHJV-UHFFFAOYSA-N | $1.9\times10^1$<br><br>4.9<br>$3.7\times10^1$<br>$5.8\times10^1$<br>$3.7\times10^1$ | | Watanabe (1993)<br><br>Kawamoto and Urano (1989)<br>HSDB (2015)<br>Woodrow et al. (1990)<br>Armbrust (2000)<br>Mackay et al. (2006d) | M<br><br>M<br>V<br>V<br>C<br>W | <br><br><br><br>12<br><br>817 |
| furosemide<br>$C_{12}H_{11}ClN_2O_5S$<br>[54-31-9]<br>ZZUFCTLCJUWOSV-UHFFFAOYSA-N | $2.5\times10^{10}$ | | HSDB (2015) | Q | 99 |
| chlorsulfuron<br>$C_{12}H_{12}ClN_5O_4S$<br>[64902-72-3]<br>VJYIFXVZLXQVHO-UHFFFAOYSA-N | $3.2\times10^4$<br>$1.5\times10^5$<br>$2.9\times10^{10}$ | | Mackay et al. (2006d)<br>Armbrust (2000)<br>Maniere et al. (2011) | V<br>C<br>? | <br><br>241, 165 |
| phosalone<br>$C_{12}H_{15}ClNO_4S_2$<br>[2310-17-0]<br>IOUNQDKNJZEDEP-UHFFFAOYSA-N | $2.5\times10^1$ | | HSDB (2015) | Q | 99 |
| dimethenamid<br>$C_{12}H_{18}ClNO_2S$<br>[87674-68-8]<br>JLYFCTQDENRSOL-UHFFFAOYSA-N | $8.9\times10^2$<br>4.7<br>$4.5\times10^2$<br>$2.3\times10^2$<br>$1.2\times10^2$ | | Keshavarz et al. (2022)<br>Duchowicz et al. (2020)<br>Hilal et al. (2008)<br>Modarresi et al. (2007)<br>Duchowicz et al. (2020) | Q<br>Q<br>Q<br>Q<br>? | <br><br><br>67<br>185, 21 |
| dimethenamid-p<br>$C_{12}H_{18}ClNO_2S$<br>[163515-14-8]<br>JLYFCTQDENRSOL-VIFPVBQESA-N | $2.1\times10^3$<br>$2.1\times10^3$ | | MacBean (2012b)<br>Maniere et al. (2011) | X<br>? | 350<br>165 |



Table A9.1: Sulfur (C, H, O, N, Cl, S) (...continued)

| Substance<br>Formula<br>(Trivial Name)<br>[CAS Registry Number]<br>InChIKey | $H_s^{cp}$<br>(at $T^\ominus$)<br>$\left[\dfrac{\mathrm{mol}}{\mathrm{m^3\,Pa}}\right]$ | $\dfrac{\mathrm{d}\ln H_s^{cp}}{\mathrm{d}(1/T)}$<br><br>[K] | Reference | Type | Note |
|---|---|---|---|---|---|
| cyazofamid<br>$C_{13}H_{13}ClN_4O_2S$<br>[120116-88-3]<br>YXKMMRDKEKCERS-UHFFFAOYSA-N | $2.5\times10^1$<br>$>2.5\times10^1$ | | HSDB (2015)<br>Maniere et al. (2011) | V<br>? | <br>241, 165 |
| metosulam<br>$C_{14}H_{13}Cl_2N_5O_4S$<br>[139528-85-1]<br>VGHPMIFEKOFHHQ-UHFFFAOYSA-N | $1.2\times10^{12}$ | | Maniere et al. (2011) | ? | 12, 165 |
| prothioconazole<br>$C_{14}H_{15}Cl_2N_3OS$<br>[178928-70-6]<br>MNHVNIJQQRJYDH-UHFFFAOYSA-N | $2.2\times10^4$<br>$>3.3\times10^4$ | | HSDB (2015)<br>Maniere et al. (2011) | V<br>? | <br>12, 165 |
| triasulfuron<br>$C_{14}H_{16}ClN_5O_5S$<br>[82097-50-5]<br>XOPFESVZMSQIKC-UHFFFAOYSA-N | $8.0\times10^8$ | | Ebert et al. (2023) | ? | 316 |
| chlorimuron-ethyl<br>$C_{15}H_{15}ClN_4O_6S$<br>[90982-32-4]<br>NSWAMPCUPHPTTC-UHFFFAOYSA-N | $5.5\times10^9$ | | HSDB (2015) | V | |
| clopidogrel<br>$C_{16}H_{16}ClNO_2S$<br>[113665-84-2]<br>GKTWGGQPFAXNFI-HNNXBMFYSA-N | $4.5\times10^3$ | | HSDB (2015) | Q | 99 |
| thenylchlor<br>$C_{16}H_{18}ClNO_2S$<br>[96491-05-3]<br>KDWQYMVPYJGPHS-UHFFFAOYSA-N | $1.2\times10^3$<br>$6.3\times10^1$ | | Duchowicz et al. (2020)<br>Duchowicz et al. (2020) | V<br>Q | 186 |
| chlorpromazine<br>$C_{17}H_{19}ClN_2S$<br>[50-53-3]<br>ZPEIMTDSQAKGNT-UHFFFAOYSA-N | $6.5\times10^4$ | | Ebert et al. (2023) | ? | 316 |
| hexythiazox<br>$C_{17}H_{21}N_2O_2ClS$<br>[78587-05-0]<br>XGWIJUOSCAQSSV-XHDPSFHLSA-N | $4.2\times10^2$<br>$8.4\times10^1$ | | HSDB (2015)<br>Maniere et al. (2011) | V<br>? | <br>12, 165 |
| clethodim<br>$C_{17}H_{26}ClNO_3S$<br>[99129-21-2]<br>INNPZTGYZSAJFN-ZTVUPKSFSA-N | $8.2\times10^5$<br>$7.1\times10^6$ | | HSDB (2015)<br>Maniere et al. (2011) | Q<br>? | 99<br>12, 165 |



Table A9.1: Sulfur (C, H, O, N, Cl, S) (...continued)

| Substance Formula (Trivial Name) [CAS Registry Number] InChIKey | $H_s^{cp}$ (at $T^\ominus$) $\left[\dfrac{\mathrm{mol}}{\mathrm{m}^3\,\mathrm{Pa}}\right]$ | $\dfrac{\mathrm{d}\ln H_s^{cp}}{\mathrm{d}(1/T)}$ [K] | Reference | Type | Note |
|---|---|---|---|---|---|
| clindamycin $C_{18}H_{33}N_2O_5ClS$ [18323-44-9] KDLRVYVGXIQJDK-AWPVFWJPSA-N | $3.4\times10^{16}$ | | HSDB (2015) | Q | 99 |
| pyridate $C_{19}H_{23}ClN_2O_2S$ [55512-33-9] JTZCTMAVMHRNTR-UHFFFAOYSA-N | $3.0\times10^2$ 4.9 $8.3\times10^3$ | | Barcelo and Hennion (1997) Goodarzi et al. (2010) Maniere et al. (2011) | X Q ? | 567 568 12, 165 |
| vismodegib $C_{19}H_{14}Cl_2N_2O_3S$ [879085-55-9] BPQMGSKTAYIVFO-UHFFFAOYSA-N | $6.2\times10^{11}$ | | HSDB (2015) | Q | 99 |
| pyridaben $C_{19}H_{25}ClN_2OS$ [96489-71-3] DWFZBUWUXWZWKD-UHFFFAOYSA-N | $2.1\times10^{-1}$ | | HSDB (2015) | V | |
| tembotrione $C_{17}H_{16}ClF_3O_6S$ [335104-84-2] IUQAXCIUEPFPSF-UHFFFAOYSA-N | $5.8\times10^9$ $5.8\times10^9$ | | HSDB (2015) Maniere et al. (2011) | V ? | 12, 165 |
| fluensulfone $C_7H_5ClF_3NO_2S_2$ [318290-98-1] XSNMWAPKHUGZGQ-UHFFFAOYSA-N | $6.1\times10^1$ | | Ebert et al. (2023) | ? | 318 |
| fluothiuron $C_{10}H_{10}Cl_2F_2N_2OS$ [33439-45-1] YFEUKKUPOVGUIW-UHFFFAOYSA-N | $>2.3\times10^{10}$ | | MacBean (2012a) | ? | |
| sulfentrazone $C_{11}H_{10}Cl_2F_2N_4O_3S$ [122836-35-5] OORLZFUTLGXMEF-UHFFFAOYSA-N | $1.5\times10^7$ $1.5\times10^7$ $3.7\times10^8$ | | Duchowicz et al. (2020) HSDB (2015) Duchowicz et al. (2020) | V V Q | 186 |
| flurazole $C_{12}H_7NO_2ClF_3S$ [72850-64-7] MKQSWTQPLLCSOB-UHFFFAOYSA-N | $4.0\times10^1$ $1.2\times10^2$ $4.0\times10^1$ | | Duchowicz et al. (2020) Duchowicz et al. (2020) MacBean (2012a) | V Q ? | 186 |
| fipronil $C_{12}H_4Cl_2F_6N_4OS$ [120068-37-3] ZOCSXAVNDGMNBV-UHFFFAOYSA-N | $1.2\times10^4$ $1.2\times10^4$ $4.4\times10^7$ | | Duchowicz et al. (2020) HSDB (2015) Duchowicz et al. (2020) | V V Q | 186 |



Table A9.1: Sulfur (C, H, O, N, Cl, S) (. . . continued)

| Substance Formula (Trivial Name) [CAS Registry Number] InChIKey | $H_s^{cp}$ (at $T^\ominus$) $\left[\dfrac{\mathrm{mol}}{\mathrm{m^3\,Pa}}\right]$ | $\dfrac{\mathrm{d}\ln H_s^{cp}}{\mathrm{d}(1/T)}$ [K] | Reference | Type | Note |
|---|---|---|---|---|---|
| flusulfamide $C_{13}H_7Cl_2F_3N_2O_4S$ [106917-52-6] GNVDAZSPJWCIQZ-UHFFFAOYSA-N | $4.0\times10^3$ | | Ebert et al. (2023) | ? | 318 |
| ethiprole $C_{13}H_9Cl_2F_3N_4OS$ [181587-01-9] FNELVJVBIYMIMC-UHFFFAOYSA-N | $2.5\times10^5$ | | Ebert et al. (2023) | ? | 318 |
| diclosulam $C_{13}H_{10}Cl_2FN_5O_3S$ [145701-21-9] QNXAVFXEJCPCJO-UHFFFAOYSA-N | $2.5\times10^{10}$ | | Ebert et al. (2023) | ? | 318 |
| cloransulam-methyl $C_{15}H_{13}ClFN_5O_5S$ [147150-35-4] BIKACRYIQSLICJ-UHFFFAOYSA-N | $1.6\times10^{11}$ | | Ebert et al. (2023) | ? | 318 |
| fluthiacet-methyl $C_{15}H_{15}ClFN_3O_3S_2$ [117337-19-6] ZCNQYNHDVRPZIH-UHFFFAOYSA-N | $4.8\times10^3$ $4.7\times10^3$ $4.2\times10^5$ $6.6\times10^5$ | | Duchowicz et al. (2020) HSDB (2015) Duchowicz et al. (2020) Modarresi et al. (2007) | V V Q Q | 186 <br> <br> <br> 67 |
| vemurafenib $C_{23}H_{18}ClF_2N_3O_3S$ [918504-65-1] GPXBXXGIAQBQNI-UHFFFAOYSA-N | $8.2\times10^{11}$ | | HSDB (2015) | Q | 99 |
| tetrabromobisphenol S $C_{12}H_6Br_4O_4S$ [39635-79-5] JHJUYGMZIWDHMO-UHFFFAOYSA-N | $1.5\times10^{11}$ $9.7\times10^5$ $5.8\times10^6$ $1.2\times10^7$ | | Zhang et al. (2010) Zhang et al. (2010) Zhang et al. (2010) Zhang et al. (2010) | Q Q Q Q | 287, 288 287, 289 287, 290 287, 291 |
| 1,3-dibromo-5-[3,5-dibromo-4-(2,3-dibromopropoxy)benzenesulfonyl]-2-(2,3-dibromopropoxy)benzene $C_{18}H_{14}Br_8O_4S$ [42757-55-1] CWZVMVIHYSYLSI-UHFFFAOYSA-N | $8.2\times10^8$ $5.2\times10^8$ $1.8\times10^{11}$ $6.4\times10^8$ | | Zhang et al. (2010) Zhang et al. (2010) Zhang et al. (2010) Zhang et al. (2010) | Q Q Q Q | 287, 288 287, 289 287, 290 287, 291 |
| bromphenol blue $C_{19}H_{10}Br_4O_5S$ [115-39-9] UDSAIICHUKSCKT-UHFFFAOYSA-N | $1.9\times10^{13}$ $5.1\times10^5$ $9.2\times10^9$ $5.3\times10^{10}$ | | Zhang et al. (2010) Zhang et al. (2010) Zhang et al. (2010) Zhang et al. (2010) | Q Q Q Q | 287, 288 287, 289 287, 290 287, 291 |



Table A9.1: Sulfur (C, H, O, N, Cl, S) (...continued)

| Substance / Formula / (Trivial Name) / [CAS Registry Number] / InChIKey | $H_s^{cp}$ (at $T^{\ominus}$) $\left[\dfrac{\mathrm{mol}}{\mathrm{m^3\,Pa}}\right]$ | $\dfrac{\mathrm{d}\ln H_s^{cp}}{\mathrm{d}(1/T)}$ [K] | Reference | Type | Note |
|---|---|---|---|---|---|
| bromcresol green $C_{21}H_{14}Br_4O_5S$ [76-60-8] FRPHFZCDPYBUAU-UHFFFAOYSA-N | $1.5\times10^{13}$ $1.0\times10^{6}$ $1.8\times10^{9}$ $1.6\times10^{10}$ | | Zhang et al. (2010) Zhang et al. (2010) Zhang et al. (2010) Zhang et al. (2010) | Q Q Q Q | 287, 288 287, 289 287, 290 287, 291 |
| bromocresol purple $C_{21}H_{16}Br_2O_5S$ [115-40-2] ABIUHPWEYMSGSR-UHFFFAOYSA-N | $9.9\times10^{12}$ | | HSDB (2015) | Q | 99 |
| difethialone $C_{31}H_{23}BrO_2S$ [104653-34-1] VSVAQRUUFVBBFS-UHFFFAOYSA-N | 9.9 9.9 $5.4\times10^{6}$ | | Duchowicz et al. (2020) HSDB (2015) Duchowicz et al. (2020) | V V Q | 186 |
| amisulbrom $C_{13}H_{13}BrFN_5O_4S_2$ [348635-87-0] BREATYVWRHIPIY-UHFFFAOYSA-N | $4.7\times10^{1}$ $3.6\times10^{4}$ | | MacBean (2012b) Maniere et al. (2011) | X ? | 350 241, 165 |
| thifluzamide $C_{13}H_6Br_2F_6N_2O_2S$ [130000-40-7] WOSNCVAPUOFXEH-UHFFFAOYSA-N | $1.2\times10^{6}$ | | Ebert et al. (2023) | ? | 318 |
| amical 48 $C_8H_8I_2O_2S$ (diiodomethyl $p$-tolyl sulfone) [20018-09-1] XOILGBPDXMVFIP-UHFFFAOYSA-N | $1.3\times10^{3}$ | | HSDB (2015) | Q | 99 |
| flubendiamide $C_{23}H_{22}F_7IN_2O_4S$ [272451-65-7] ZGNITFSDLCMLGI-UHFFFAOYSA-N | $4.5\times10^{-2}$ | | HSDB (2015) | V | |



## A10  Organic species with phosphorus (P)

### A10.1  Phosphorus (C, H, O, N, Cl, Br, S, P)

Table A10.1: Phosphorus (C, H, O, N, Cl, Br, S, P)

| Substance Formula (Trivial Name) [CAS Registry Number] InChIKey | $H_s^{cp}$ (at $T^\ominus$) $\left[\dfrac{\mathrm{mol}}{\mathrm{m}^3\,\mathrm{Pa}}\right]$ | $\dfrac{\mathrm{d}\ln H_s^{cp}}{\mathrm{d}(1/T)}$ [K] | Reference | Type | Note |
|---|---|---|---|---|---|
| 9-icosyl-9-phosphabicyclo[4.2.1]nonane | $3.1\times10^{-5}$ | | Zhang et al. (2010) | Q | 287, 288 |
| $C_{28}H_{55}P$ | $3.1\times10^{-3}$ | | Zhang et al. (2010) | Q | 287, 289 |
| [13886-99-2] | $2.2\times10^{-2}$ | | Zhang et al. (2010) | Q | 287, 290 |
| UNOOEFGBOLKBFW-UHFFFAOYSA-N | $8.0\times10^{-6}$ | | Zhang et al. (2010) | Q | 287, 291 |
| triphenylphosphine | $4.3\times10^{2}$ | | Zhang et al. (2010) | Q | 287, 288 |
| $C_{18}H_{15}P$ | $9.5\times10^{-3}$ | | Zhang et al. (2010) | Q | 287, 289 |
| [603-35-0] | $1.3\times10^{1}$ | | Zhang et al. (2010) | Q | 287, 290 |
| RIOQSEWOXXDEQQ-UHFFFAOYSA-N | 4.8 | | Zhang et al. (2010) | Q | 287, 291 |
| phosphoric acid, trimethyl ester | $1.4\times10^{3}$ | | Wolfenden and Williams (1983) | M | 12 |
| $C_3H_9O_4P$ | $6.3\times10^{1}$ | | Yaws (2003) | X | 237 |
| (trimethyl phosphate) | $6.1\times10^{1}$ | | Gharagheizi et al. (2010) | Q | 246 |
| [512-56-1] | $1.4\times10^{3}$ | | Bartelt-Hunt et al. (2008) | ? | 21 |
| WVLBCYQITXONBZ-UHFFFAOYSA-N | $5.2\times10^{1}$ | | Yaws (1999) | ? | 21 |
| trimethyl phosphite | $9.0\times10^{-1}$ | | HSDB (2015) | Q | 99 |
| $C_3H_9O_3P$ | | | | | |
| [121-45-9] | | | | | |
| CYTQBVOFDCPGCX-UHFFFAOYSA-N | | | | | |
| dimethyl methylphosphonate | 7.6 | | HSDB (2015) | Q | 99 |
| $C_3H_9O_3P$ | 7.6 | | Bartelt-Hunt et al. (2008) | ? | 21 |
| [756-79-6] | | | | | |
| VONWDASPFIQPDY-UHFFFAOYSA-N | | | | | |
| diethyl hydrogen phosphite | 1.7 | | HSDB (2015) | Q | 99 |
| $C_4H_{11}O_3P$ | $5.5\times10^{-3}$ | | Bartelt-Hunt et al. (2008) | ? | 21 |
| [762-04-9] | | | | | |
| MJUJXFBTEFXVKU-UHFFFAOYSA-N | | | | | |
| triethylphosphate | $2.7\times10^{2}$ | | Wolfenden and Williams (1983) | M | 12 |
| $C_6H_{15}O_4P$ | $1.4\times10^{2}$ | | Abraham et al. (1994a) | R | |
| [78-40-0] | $6.7\times10^{1}$ | | Yaws (2003) | X | 258 |
| DQWPFSLDHJDLRL-UHFFFAOYSA-N | $6.6\times10^{1}$ | | Yaws (2003) | X | 237 |
| | $5.2\times10^{1}$ | | Dupeux et al. (2022) | Q | 259 |
| | $6.7\times10^{1}$ | | Gharagheizi et al. (2010) | Q | 246 |
| | 6.6 | | Bartelt-Hunt et al. (2008) | ? | 21 |
| | $9.6\times10^{1}$ | | Yaws (1999) | ? | 21 |
| diethyl ethyl phosphonate | 3.4 | | Bartelt-Hunt et al. (2008) | ? | 21 |
| $C_6H_{15}O_3P$ | | | | | |
| [78-38-6] | | | | | |
| AATNZNJRDOVKDD-UHFFFAOYSA-N | | | | | |





Table A10.1: Phosphorus (C, H, O, N, Cl, Br, S, P) (...continued)

| Substance Formula (Trivial Name) [CAS Registry Number] InChIKey | $H_s^{cp}$ (at $T^\ominus$) $\left[\dfrac{\text{mol}}{\text{m}^3\,\text{Pa}}\right]$ | $\dfrac{\text{d}\ln H_s^{cp}}{\text{d}(1/T)}$ [K] | Reference | Type | Note |
|---|---|---|---|---|---|
| mevinphos $C_7H_{13}O_6P$ [7786-34-7] GEPDYQSQVLXLEU-AATRIKPKSA-N | $2.4\times10^5$ $2.5\times10^3$ | | Mackay et al. (2006d) Sanders and Seiber (1983) HSDB (2015) | V V Q | 558 87 99 |
| diisopropyl methanephosphonate $C_7H_{17}O_3P$ [1445-75-6] WOAFDHWYKSOANX-UHFFFAOYSA-N | $2.2\times10^{-1}$ $2.2\times10^{-1}$ | | HSDB (2015) Bartelt-Hunt et al. (2008) | V ? | 21 |
| dibutyl hydrogen phosphite $C_8H_{19}O_3P$ [1809-19-4] NFJPGAKRJKLOJK-UHFFFAOYSA-N | $5.5\times10^{-1}$ | | HSDB (2015) | Q | 99 |
| dibutyl phosphate $C_8H_{19}O_4P$ [107-66-4] JYFHYPJRHGVZDY-UHFFFAOYSA-N | $2.3\times10^3$ | | HSDB (2015) | Q | 99 |
| tetraethyl pyrophosphate $C_8H_{20}O_7P_2$ [107-49-3] IDCBOTIENDVCBQ-UHFFFAOYSA-N | $4.5\times10^4$ | | HSDB (2015) | V | |
| tripropyl phosphate $C_9H_{21}O_4P$ [513-08-6] RXPQRKFMDQNODS-UHFFFAOYSA-N | $1.5\times10^1$ | | Wolfenden and Williams (1983) | M | 12 |
| triallyl phosphate $C_9H_{15}O_4P$ [1623-19-4] XHGIFBQQEGRTPB-UHFFFAOYSA-N | $1.8\times10^1$ | | HSDB (2015) | Q | 99 |
| tributylphosphate $C_{12}H_{27}O_4P$ [126-73-8] STCOOQWBFONSKY-UHFFFAOYSA-N | $7.0$ $1.6\times10^1$ $4.8$ | | HSDB (2015) Glotfelty et al. (1987) Yoshida et al. (1983) | V V V | |
| hexaethyl tetraphosphate $C_{12}H_{30}O_{13}P_4$ [757-58-4] DAJYZXUXDOSMCG-UHFFFAOYSA-N | $3.0\times10^{11}$ | | HSDB (2015) | Q | 99 |
| crotoxyphos $C_{14}H_{19}O_6P$ [7700-17-6] XXXSILNSXNPGKG-ZHACJKMWSA-N | $1.7\times10^3$ $1.7\times10^3$ $1.7\times10^3$ | | HSDB (2015) Mackay et al. (2006d) MacBean (2012a) | V V ? | |





Table A10.1: Phosphorus (C, H, O, N, Cl, Br, S, P) (...continued)

| Substance<br>Formula<br>(Trivial Name)<br>[CAS Registry Number]<br>InChIKey | $H_s^{cp}$<br>(at $T^{\ominus}$)<br>$\left[\dfrac{\text{mol}}{\text{m}^3\,\text{Pa}}\right]$ | $\dfrac{\text{d}\ln H_s^{cp}}{\text{d}(1/T)}$<br><br>[K] | Reference | Type | Note |
|---|---|---|---|---|---|
| phosphoric acid, dibutyl phenyl ester<br>$C_{14}H_{23}O_4P$<br>[2528-36-1]<br>YICSVBJRVMLQNS-UHFFFAOYSA-N | $2.0\times10^1$ | | HSDB (2015) | Q | 99 |
| bis(2-ethylhexyl) hydrogen phosphite<br>$C_{16}H_{35}O_3P$<br>[3658-48-8]<br>HZIUHEQKVCPTAJ-UHFFFAOYSA-N | $5.8\times10^{-2}$<br>$6.6\times10^{-4}$ | | HSDB (2015)<br>Bartelt-Hunt et al. (2008) | Q<br>? | 99<br>21 |
| bis(2-ethylhexyl)hydrogen phosphate<br>$C_{16}H_{35}O_4P$<br>(bis(2-ethylhexyl) phosphate)<br>[298-07-7]<br>SEGLCEQVOFDUPX-UHFFFAOYSA-N | $2.4\times10^2$ | | HSDB (2015) | Q | 99 |
| triphenyl phosphate<br>$C_{18}H_{15}O_4P$<br>[115-86-6]<br>XZZNDPSIHUTMOC-UHFFFAOYSA-N | $3.0$ | | HSDB (2015) | V | |
| tris(2-butoxyethyl) phosphate<br>$C_{18}H_{39}O_7P$<br>[78-51-3]<br>WTLBZVNBAKMVDP-UHFFFAOYSA-N | $8.2\times10^5$ | | HSDB (2015) | Q | 99 |
| $p$-cresyl diphenyl phosphate<br>$C_{19}H_{17}O_4P$<br>[78-31-9]<br>OJUZRFGUKHQNJX-UHFFFAOYSA-N | $9.9\times10^1$ | | HSDB (2015) | Q | 447 |
| triphenylphosphine oxide<br>$C_{18}H_{15}OP$<br>[791-28-6]<br>FIQMHBFVRAXMOP-UHFFFAOYSA-N | $1.9\times10^4$<br>$4.6\times10^4$<br>$1.1\times10^7$<br>$2.5\times10^{-1}$ | | Zhang et al. (2010)<br>Zhang et al. (2010)<br>Zhang et al. (2010)<br>Zhang et al. (2010) | Q<br>Q<br>Q<br>Q | 287, 288<br>287, 289<br>287, 290<br>287, 291 |
| phosphorous acid, triphenyl ester<br>$C_{18}H_{15}O_3P$<br>[101-02-0]<br>HVLLSGMXQDNUAL-UHFFFAOYSA-N | $1.8\times10^1$<br>$4.4\times10^{-2}$<br>$1.5\times10^2$<br>$7.0\times10^4$ | | Zhang et al. (2010)<br>Zhang et al. (2010)<br>Zhang et al. (2010)<br>Zhang et al. (2010) | Q<br>Q<br>Q<br>Q | 287, 288<br>287, 289<br>287, 290<br>287, 291 |
| trihexylphosphine oxide<br>$C_{18}H_{39}OP$<br>[3084-48-8]<br>PPDZLUVUQQGIOJ-UHFFFAOYSA-N | $4.5\times10^{-3}$<br>$2.9\times10^{-3}$<br>$5.8\times10^4$<br>$3.5\times10^{-7}$ | | Zhang et al. (2010)<br>Zhang et al. (2010)<br>Zhang et al. (2010)<br>Zhang et al. (2010) | Q<br>Q<br>Q<br>Q | 287, 288<br>287, 289<br>287, 290<br>287, 291 |



Table A10.1: Phosphorus (C, H, O, N, Cl, Br, S, P) (...continued)

| Substance Formula (Trivial Name) [CAS Registry Number] InChIKey | $H_s^{cp}$ (at $T^\ominus$) $\left[\dfrac{\text{mol}}{\text{m}^3\,\text{Pa}}\right]$ | $\dfrac{\text{d}\ln H_s^{cp}}{\text{d}(1/T)}$ [K] | Reference | Type | Note |
|---|---|---|---|---|---|
| phosphoric acid, octyl diphenyl ester C$_{20}$H$_{27}$O$_4$P [115-88-8] YAFOVCNAQTZDQB-UHFFFAOYSA-N | $3.9\times10^1$ | | HSDB (2015) | Q | 99 |
| octyldihexylphosphine oxide C$_{20}$H$_{43}$OP [31160-64-2] XHRRUIJGMKIISX-UHFFFAOYSA-N | $2.5\times10^{-3}$ $3.1\times10^{-3}$ $5.3\times10^4$ $2.3\times10^{-7}$ | | Zhang et al. (2010) Zhang et al. (2010) Zhang et al. (2010) Zhang et al. (2010) | Q Q Q Q | 287, 288 287, 289 287, 290 287, 291 |
| octicizer C$_{20}$H$_{27}$O$_4$P [1241-94-7] CGSLYBDCEGBZCG-UHFFFAOYSA-N | $1.0\times10^{-1}$ 2.7 $1.8\times10^{-1}$ | | Keshavarz et al. (2022) Duchowicz et al. (2020) Duchowicz et al. (2020) | Q Q ? | 185, 21 |
| tris(methylphenyl) phosphate C$_{21}$H$_{21}$O$_4$P (tricresyl phosphate) [1330-78-5] IUJIYUAKFBGBCG-UHFFFAOYSA-N | $1.2\times10^1$ | | HSDB (2015) | V | |
| phosphoric acid, (1-methylethyl)phenyl diphenyl ester C$_{21}$H$_{21}$O$_4$P [28108-99-8] JJXNVYMIYBNZQX-UHFFFAOYSA-N | $1.3\times10^2$ | | HSDB (2015) | Q | 99 |
| phosphoric acid, tris(2-methylphenyl) ester C$_{21}$H$_{21}$O$_4$P (tri-$o$-cresyl phosphate) [78-30-8] YSMRWXYRXBRSND-UHFFFAOYSA-N | 5.2 | | HSDB (2015) | Q | 447 |
| phosphoric acid, tris(3-methylphenyl) ester C$_{21}$H$_{21}$O$_4$P (tri-$m$-cresyl phosphate) [563-04-2] RMLPZKRPSQVRAB-UHFFFAOYSA-N | $1.4\times10^{-1}$ 5.1 9.9 $1.2\times10^{-1}$ | | Keshavarz et al. (2022) Duchowicz et al. (2020) HSDB (2015) Duchowicz et al. (2020) | Q Q Q ? | 447 185, 21 |
| phosphoric acid, tris(4-methylphenyl) ester C$_{21}$H$_{21}$O$_4$P (tri-$p$-cresyl phosphate) [78-32-0] BOSMZFBHAYFUBJ-UHFFFAOYSA-N | $1.8\times10^2$ | | HSDB (2015) | Q | 447 |




Table A10.1: Phosphorus (C, H, O, N, Cl, Br, S, P) (...continued)

| Substance<br>Formula<br>(Trivial Name)<br>[CAS Registry Number]<br>InChIKey | $H_s^{cp}$<br>(at $T^{\ominus}$)<br>$\left[\dfrac{\mathrm{mol}}{\mathrm{m^3\,Pa}}\right]$ | $\dfrac{\mathrm{d}\ln H_s^{cp}}{\mathrm{d}(1/T)}$<br><br>[K] | Reference | Type | Note |
|---|---|---|---|---|---|
| (3-*tert*-butylphenyl) diphenyl phosphate<br>$C_{22}H_{23}O_4P$<br>NIAVXHWAURFNOW-UHFFFAOYSA-N | $1.1\times10^1$ | | Ebert et al. (2023) | ? | 365 |
| (4-*tert*-butylphenyl) diphenyl phosphate<br>$C_{22}H_{23}O_4P$<br>[981-40-8]<br>ULGAVXUJJBOWOD-UHFFFAOYSA-N | $4.5\times10^1$<br>$1.8\times10^2$<br>3.6<br>$1.1\times10^1$ | | HSDB (2015)<br>Keshavarz et al. (2022)<br>Duchowicz et al. (2020)<br>Duchowicz et al. (2020) | V<br>Q<br>Q<br>? | <br><br><br>185, 21 |
| isodecyl diphenyl phosphate<br>$C_{22}H_{31}O_4P$<br>[29761-21-5]<br>RYUJRXVZSJCHDZ-UHFFFAOYSA-N | $2.3\times10^1$ | | HSDB (2015) | Q | 99 |
| dioctylhexylphosphine oxide<br>$C_{22}H_{47}OP$<br>[31160-66-4]<br>MKEFGIKZZDCMQC-UHFFFAOYSA-N | $1.4\times10^{-3}$<br>$3.4\times10^{-3}$<br>$4.0\times10^4$<br>$1.4\times10^{-7}$ | | Zhang et al. (2010)<br>Zhang et al. (2010)<br>Zhang et al. (2010)<br>Zhang et al. (2010) | Q<br>Q<br>Q<br>Q | 287, 288<br>287, 289<br>287, 290<br>287, 291 |
| tris(2,4-dimethylphenyl)phosphate<br>$C_{24}H_{27}O_4P$<br>[3862-12-2]<br>KOWVWXQNQNCRRS-UHFFFAOYSA-N | $1.4\times10^2$ | | HSDB (2015) | Q | 99 |
| tris(2,5-dimethylphenyl)phosphate<br>$C_{24}H_{27}O_4P$<br>[19074-59-0]<br>MDHAARLWBHZGIP-UHFFFAOYSA-N | $1.4\times10^2$ | | HSDB (2015) | Q | 99 |
| tris(2,6-dimethylphenyl)phosphate<br>$C_{24}H_{27}O_4P$<br>[121-06-2]<br>QLORRTLBSJTMSN-UHFFFAOYSA-N | $1.4\times10^2$ | | HSDB (2015) | Q | 99 |
| tris(3,4-dimethylphenyl)phosphate<br>$C_{24}H_{27}O_4P$<br>[3862-11-1]<br>BCTKCHOESSAGCN-UHFFFAOYSA-N | $1.4\times10^2$ | | HSDB (2015) | Q | 99 |
| tris(3,5-dimethylphenyl)phosphate<br>$C_{24}H_{27}O_4P$<br>[25653-16-1]<br>LLPMAOBOEQFPRE-UHFFFAOYSA-N | $1.4\times10^2$ | | HSDB (2015) | Q | 99 |
| tris(4-isopropylphenyl) phosphate<br>$C_{27}H_{33}O_4P$<br>[26967-76-0]<br>ANVREEJNGJMLOV-UHFFFAOYSA-N | $3.4\times10^1$ | | HSDB (2015) | Q | 99 |



Table A10.1: Phosphorus (C, H, O, N, Cl, Br, S, P) (... continued)

| Substance<br>Formula<br>(Trivial Name)<br>[CAS Registry Number]<br>InChIKey | $H_s^{cp}$ (at $T^{\ominus}$) $\left[\dfrac{\mathrm{mol}}{\mathrm{m}^3\,\mathrm{Pa}}\right]$ | $\dfrac{\mathrm{d}\ln H_s^{cp}}{\mathrm{d}(1/T)}$ [K] | Reference | Type | Note |
|---|---|---|---|---|---|
| trioctylphosphine oxide | $8.2\times10^{-4}$ | | Zhang et al. (2010) | Q | 287, 288 |
| $C_{24}H_{51}OP$ | $3.7\times10^{-3}$ | | Zhang et al. (2010) | Q | 287, 289 |
| [78-50-2] | $3.4\times10^{4}$ | | Zhang et al. (2010) | Q | 287, 290 |
| ZMBHCYHQLYEYDV-UHFFFAOYSA-N | $9.2\times10^{-8}$ | | Zhang et al. (2010) | Q | 287, 291 |
| bis(2-ethylhexyl)-2-ethylhexyl | $2.1\times10^{-2}$ | | Zhang et al. (2010) | Q | 287, 288 |
| phosphonate | | | | | |
| $C_{24}H_{51}O_3P$ | $6.2\times10^{-6}$ | | Zhang et al. (2010) | Q | 287, 289 |
| [126-63-6] | $5.4\times10^{4}$ | | Zhang et al. (2010) | Q | 287, 290 |
| GOCVCBDBQYEFQD-UHFFFAOYSA-N | $7.7\times10^{-5}$ | | Zhang et al. (2010) | Q | 287, 291 |
| | $1.3\times10^{-3}$ | | Bartelt-Hunt et al. (2008) | ? | 21 |
| didodecyl hydrogen phosphate | $2.5\times10^{1}$ | | HSDB (2015) | Q | 99 |
| $C_{24}H_{51}O_4P$ | | | | | |
| [7057-92-3] | | | | | |
| JTXUVYOABGUBMX-UHFFFAOYSA-N | | | | | |
| phosphoric acid, tris(2-ethylhexyl) | $1.2\times10^{2}$ | | HSDB (2015) | V | |
| ester | | | | | |
| $C_{24}H_{51}O_4P$ | | | | | |
| (trioctyl phosphate) | | | | | |
| [78-42-2] | | | | | |
| GTVWRXDRKAHEAD-UHFFFAOYSA-N | | | | | |
| diisodecylphenyl phosphite | 1.9 | | Zhang et al. (2010) | Q | 287, 288 |
| $C_{26}H_{47}O_3P$ | 2.5 | | Zhang et al. (2010) | Q | 287, 289 |
| [25550-98-5] | $1.6\times10^{3}$ | | Zhang et al. (2010) | Q | 287, 290 |
| SXXILWLQSQDLDL-UHFFFAOYSA-N | $3.9\times10^{-3}$ | | Zhang et al. (2010) | Q | 287, 291 |
| 4-nonylphenyl diphenyl phosphate | $8.1\times10^{2}$ | | Keshavarz et al. (2022) | Q | |
| $C_{27}H_{33}O_4P$ | $2.5\times10^{1}$ | | Duchowicz et al. (2020) | Q | 184 |
| [64532-97-4] | $7.0\times10^{2}$ | | Duchowicz et al. (2020) | ? | 185, 21 |
| LMCLPMXCYFSRNG-UHFFFAOYSA-N | | | | | |
| resorcinol bis(diphenyl phosphate) | $3.4\times10^{7}$ | | Zhang et al. (2010) | Q | 287, 288 |
| $C_{30}H_{24}O_8P_2$ | $1.4\times10^{-2}$ | | Zhang et al. (2010) | Q | 287, 289 |
| [57583-54-7] | $2.6\times10^{8}$ | | Zhang et al. (2010) | Q | 287, 290 |
| OWICEWMBIBPFAH-UHFFFAOYSA-N | $3.9\times10^{16}$ | | Zhang et al. (2010) | Q | 287, 291 |
| hydroquinone bis(diphenyl | $3.4\times10^{12}$ | | Abraham et al. (2019) | Q | |
| phosphate) | | | | | |
| $C_{30}H_{24}O_8P_2$ | | | | | |
| [51732-57-1] | | | | | |
| RECLNCPBBUHRDY-UHFFFAOYSA-N | | | | | |
| tris(4-*tert*-butylphenyl) phosphate | $1.4\times10^{1}$ | | Zhang et al. (2010) | Q | 287, 288 |
| $C_{30}H_{39}O_4P$ | $8.4\times10^{-4}$ | | Zhang et al. (2010) | Q | 287, 289 |
| [78-33-1] | $1.6\times10^{3}$ | | Zhang et al. (2010) | Q | 287, 290 |
| LORSVOJSXMHDHF-UHFFFAOYSA-N | $3.5\times10^{1}$ | | Zhang et al. (2010) | Q | 287, 291 |



Table A10.1: Phosphorus (C, H, O, N, Cl, Br, S, P) (...continued)

| Substance Formula (Trivial Name) [CAS Registry Number] InChIKey | $H_s^{cp}$ (at $T^{\ominus}$) $\left[\dfrac{\mathrm{mol}}{\mathrm{m^3\,Pa}}\right]$ | $\dfrac{\mathrm{d}\ln H_s^{cp}}{\mathrm{d}(1/T)}$ [K] | Reference | Type | Note |
|---|---|---|---|---|---|
| tris-(2,4-di-*tert*-butylphenyl) phosphite | $6.1\times10^{-2}$ | | Zhang et al. (2010) | Q | 287, 288 |
| $C_{42}H_{63}O_3P$ | $6.5\times10^{-5}$ | | Zhang et al. (2010) | Q | 287, 289 |
| [31570-04-4] | $1.5\times10^2$ | | Zhang et al. (2010) | Q | 287, 290 |
| JKIJEFPNVSHHEI-UHFFFAOYSA-N | 5.8 | | Zhang et al. (2010) | Q | 287, 291 |
| glyphosate | $1.8\times10^6$ | | Mackay et al. (2006d) | V | |
| $C_3H_8NO_5P$ | $4.8\times10^6$ | | Maniere et al. (2011) | ? | 165 |
| [1071-83-6] | | | | | |
| XDDAORKBJWWYJS-UHFFFAOYSA-N | | | | | |
| krenite | $2.0\times10^7$ | | HSDB (2015) | V | |
| $C_3H_{11}N_2O_4P$ | | | | | |
| (fosamine-ammonium) | | | | | |
| [25954-13-6] | | | | | |
| OTSAMNSACVKIOJ-UHFFFAOYSA-N | | | | | |
| tabun | $6.6\times10^1$ | | HSDB (2015) | V | |
| $C_5H_{11}N_2O_2P$ | $6.2\times10^2$ | | Bartelt-Hunt et al. (2008) | ? | 21 |
| [77-81-6] | $6.5\times10^1$ | | Opresko et al. (1998) | ? | |
| PJVJTCIRVMBVIA-UHFFFAOYSA-N | | | | | |
| glufosinate-ammonium | $2.2\times10^8$ | | MacBean (2012b) | X | 350 |
| $C_5H_{15}N_2O_4P$ | | | | | |
| [77182-82-2] | | | | | |
| ZBMRKNMTMPPMMK-UHFFFAOYSA-N | | | | | |
| hexamethylphosphoramide | $4.9\times10^2$ | | Duchowicz et al. (2020) | V | 186 |
| $C_6H_{18}N_3OP$ | $5.3\times10^{-4}$ | | Duchowicz et al. (2020) | Q | |
| [680-31-9] | | | | | |
| GNOIPBMMFNIUFM-UHFFFAOYSA-N | | | | | |
| monocrotophos | $1.5\times10^7$ | | HSDB (2015) | V | |
| $C_7H_{14}NO_5P$ | | | Mackay et al. (2006d) | V | 558 |
| [6923-22-4] | | | | | |
| KRTSDMXIXPKRQR-AATRIKPKSA-N | | | | | |
| dicrotophos | $2.0\times10^5$ | | Mackay et al. (2006d) | V | |
| $C_8H_{16}NO_5P$ | $7.8\times10^4$ | | Keshavarz et al. (2022) | Q | |
| [141-66-2] | $1.0\times10^3$ | | Duchowicz et al. (2020) | Q | |
| VEENJGZXVHKXNB-VOTSOKGWSA-N | $2.0\times10^5$ | | Duchowicz et al. (2020) | ? | 185, 21 |
| octamethyldiphosphoramide | $1.6\times10^{11}$ | | HSDB (2015) | Q | 99 |
| $C_8H_{24}N_4O_3P_2$ | | | | | |
| (schradan) | | | | | |
| [152-16-9] | | | | | |
| SZKKRCSOSQAJDE-UHFFFAOYSA-N | | | | | |





Table A10.1: Phosphorus (C, H, O, N, Cl, Br, S, P) (...continued)

| Substance<br>Formula<br>(Trivial Name)<br>[CAS Registry Number]<br>InChIKey | $H_s^{cp}$<br>(at $T^\ominus$)<br><br>$\left[\dfrac{\text{mol}}{\text{m}^3\,\text{Pa}}\right]$ | $\dfrac{\text{d}\ln H_s^{cp}}{\text{d}(1/T)}$<br><br>[K] | Reference | Type | Note |
|---|---|---|---|---|---|
| fyrol 6<br>$C_9H_{22}NO_5P$<br>(diethyl ((diethanolamino)methyl)<br>phosphonate)<br>[2781-11-5]<br>CCJKFLLIJCGHMO-UHFFFAOYSA-N | $6.2\times10^1$ | | HSDB (2015) | V | |
| diethyl 4-nitrophenyl phosphate<br>$C_{10}H_{14}NO_6P$<br>(paraoxon)<br>[311-45-5]<br>WYMSBXTXOHUIGT-UHFFFAOYSA-N | $9.1\times10^4$<br>$1.6\times10^3$<br>$9.0\times10^1$<br>$1.5\times10^4$<br>$2.7\times10^{-1}$ | | Duchowicz et al. (2020)<br>Glotfelty et al. (1987)<br>Duchowicz et al. (2020)<br>HSDB (2015)<br>Bartelt-Hunt et al. (2008) | V<br>V<br>Q<br>Q<br>? | 186<br><br><br>99<br>21 |
| dimethyl 4-nitrophenyl phosphate<br>$C_8H_{10}NO_6P$<br>(methyl paraoxon)<br>[950-35-6]<br>BAFQDKPJKOLXFZ-UHFFFAOYSA-N | $>1.1\times10^4$ | | Woodrow et al. (1990) | V | |
| buminafos<br>$C_{18}H_{38}NO_3P$<br>[51249-05-9]<br>NMBXMBCZBXUXAM-UHFFFAOYSA-N | 5.0 | | MacBean (2012a) | ? | 12 |
| methylphosphonyldifluoride<br>$CH_3F_2OP$<br>[676-99-3]<br>PQIOSYKVBBWRRI-UHFFFAOYSA-N | $4.5\times10^{-1}$ | | HSDB (2015) | Q | 99 |
| sarin<br>$C_4H_{10}FO_2P$<br>[107-44-8]<br>DYAHQFWOVKZOOW-UHFFFAOYSA-N | $1.7\times10^1$<br>1.1<br>$1.8\times10^1$ | | HSDB (2015)<br>Bartelt-Hunt et al. (2008)<br>Opresko et al. (1998) | V<br>?<br>? | <br>21<br> |
| dimefox<br>$C_4H_{12}FN_2OP$<br>[115-26-4]<br>PGJBQBDNXAZHBP-UHFFFAOYSA-N | $4.5\times10^2$ | | HSDB (2015) | V | |
| isoflurophate<br>$C_6H_{14}FO_3P$<br>(diisopropyl fluorophosphate)<br>[55-91-4]<br>MUCZHBLJLSDCSD-UHFFFAOYSA-N | 3.1<br>$7.6\times10^{-5}$ | | HSDB (2015)<br>Bartelt-Hunt et al. (2008) | Q<br>? | 99<br>21 |
| cyclohexyl<br>methylphosphonofluoridate<br>$C_7H_{14}FO_2P$<br>(cyclosarin)<br>[329-99-7]<br>SNTRKUOVAPUGAY-UHFFFAOYSA-N | 3.5<br><br>3.5<br>$4.3\times10^3$ | | Duchowicz et al. (2020)<br><br>HSDB (2015)<br>Duchowicz et al. (2020) | V<br><br>V<br>Q | 186<br><br><br> |



Table A10.1: Phosphorus (C, H, O, N, Cl, Br, S, P) (...continued)

| Substance<br>Formula<br>(Trivial Name)<br>[CAS Registry Number]<br>InChIKey | $H_s^{cp}$<br>(at $T^{\ominus}$)<br>$\left[\dfrac{\text{mol}}{\text{m}^3\,\text{Pa}}\right]$ | $\dfrac{\text{d}\ln H_s^{cp}}{\text{d}(1/T)}$<br><br>[K] | Reference | Type | Note |
|---|---|---|---|---|---|
| soman<br>$C_7H_{16}FO_2P$<br>[96-64-0]<br>GRXKLBBBQUKJJZ-UHFFFAOYSA-N | 2.1<br>2.1<br>2.2 | | HSDB (2015)<br>Bartelt-Hunt et al. (2008)<br>Opresko et al. (1998) | V<br>?<br>? | <br>21<br> |
| mipafox<br>$C_6H_{16}FN_2OP$<br>[371-86-8]<br>UOSHUBFBCPGQAY-UHFFFAOYSA-N | $3.3\times10^3$ | | HSDB (2015) | V | |
| phenylphosphonous dichloride<br>$C_6H_5Cl_2P$<br>[644-97-3]<br>IMDXZWRLUZPMDH-UHFFFAOYSA-N | $6.5\times10^{-1}$<br>$2.5\times10^{-3}$<br>$6.2\times10^{-2}$<br>$3.3\times10^{-2}$ | | Zhang et al. (2010)<br>Zhang et al. (2010)<br>Zhang et al. (2010)<br>Zhang et al. (2010) | Q<br>Q<br>Q<br>Q | 287, 288<br>287, 289<br>287, 290<br>287, 291 |
| chlorphonium chloride<br>$C_{19}H_{32}Cl_3P$<br>[115-78-6]<br>IVHVNMLJNASKHW-UHFFFAOYSA-M | $2.8\times10^7$ | | MacBean (2012a) | ? | 12 |
| triclofos<br>$C_2H_4Cl_3O_4P$<br>[306-52-5]<br>YYQRGCZGSFRBAM-UHFFFAOYSA-N | $7.0\times10^7$ | | HSDB (2015) | Q | 99 |
| (2-chloroethyl)-phosphonic acid<br>$C_2H_6ClO_3P$<br>(ethephon)<br>[16672-87-0]<br>UDPGUMQDCGORJQ-UHFFFAOYSA-N | $6.9\times10^7$<br>$>6.9\times10^6$ | | HSDB (2015)<br>Maniere et al. (2011) | V<br>? | <br>241, 165 |
| 1-hydroxy-2,2,2-<br>trichloroethylphosphonic acid,<br>dimethyl ester<br>$C_4H_8Cl_3O_4P$<br>(trichlorfon)<br>[52-68-6]<br>NFACJZMKEDPNKN-UHFFFAOYSA-N | $5.5\times10^1$<br><br><br>$>8.1\times10^2$<br>$5.8\times10^5$<br>$6.0\times10^5$<br>$5.9\times10^5$<br>$5.8\times10^3$<br>$3.8\times10^1$ | | Chao et al. (2017)<br><br><br>Kawamoto and Urano (1989)<br>HSDB (2015)<br>Mackay et al. (2006d)<br>Suntio et al. (1988)<br>Barcelo and Hennion (1997)<br>Goodarzi et al. (2010) | M<br><br><br>M<br>V<br>V<br>V<br>X<br>Q | <br><br><br><br><br><br>12<br>567<br>568 |
| dimethyl-2,2-dichlorovinyl<br>phosphate<br>$C_4H_7Cl_2O_4P$<br>(dichlorvos)<br>[62-73-7]<br>OEBRKCOSUFCWJD-UHFFFAOYSA-N | $3.9\times10^1$<br><br>$8.1\times10^{-2}$<br>$1.7\times10^1$<br>5.2<br>5.3<br>$5.2\times10^{-2}$<br>$9.7\times10^{-1}$ | 11000 | Gautier et al. (2003)<br><br>Kawamoto and Urano (1989)<br>HSDB (2015)<br>Mackay et al. (2006d)<br>Suntio et al. (1988)<br>Barcelo and Hennion (1997)<br>Goodarzi et al. (2010) | M<br><br>M<br>V<br>V<br>V<br>X<br>Q | <br><br><br><br><br>12<br>567<br>568, 571 |



Table A10.1: Phosphorus (C, H, O, N, Cl, Br, S, P) (...continued)

| Substance<br>Formula<br>(Trivial Name)<br>[CAS Registry Number]<br>InChIKey | $H_s^{cp}$<br>(at $T^\ominus$)<br>$\left[\dfrac{\text{mol}}{\text{m}^3\,\text{Pa}}\right]$ | $\dfrac{\text{d}\ln H_s^{cp}}{\text{d}(1/T)}$<br><br>[K] | Reference | Type | Note |
|---|---|---|---|---|---|
| tris(2-chloroethyl) phosphate<br>$C_6H_{12}Cl_3O_4P$<br>[115-96-8]<br>HQUQLFOMPYWACS-UHFFFAOYSA-N | 3.0 | | HSDB (2015) | V | |
| cyclophosphamide<br>$C_7H_{15}Cl_2N_2O_2P$<br>[50-18-0]<br>CMSMOCZEIVJLDB-UHFFFAOYSA-N | $7.0\times10^5$ | | HSDB (2015) | Q | 99 |
| ifosfamide<br>$C_7H_{15}Cl_2N_2O_2P$<br>[3778-73-2]<br>HOMGKSMUEGBAAB-UHFFFAOYSA-N | $7.0\times10^5$ | | HSDB (2015) | Q | 99 |
| butonate<br>$C_8H_{14}Cl_3O_5P$<br>[126-22-7]<br>BKAQXYNWONVOAX-UHFFFAOYSA-N | $3.3\times10^4$ | | HSDB (2015) | Q | 99 |
| phosphoric acid,<br>7-chlorobicyclo[3.2.0]hepta-2,6-<br>dien-6-yl dimethyl<br>ester<br>$C_9H_{12}ClO_4P$<br>(heptenophos)<br>[23560-59-0]<br>GBAWQJNHVWMTLU-UHFFFAOYSA-N | $5.8\times10^1$<br><br><br><br>$4.3\times10^3$ | | HSDB (2015)<br><br><br><br>MacBean (2012a) | V<br><br><br><br>? | |
| tris(2,3-dichloropropyl) phosphate<br>$C_9H_{15}Cl_6O_4P$<br>[78-43-3]<br>JZZBTMVTLBHJHL-UHFFFAOYSA-N | $3.8\times10^3$ | | HSDB (2015) | Q | 99 |
| tris(1,3-<br>dichloroisopropyl)phosphate<br>$C_9H_{15}Cl_6O_4P$<br>[13674-87-8]<br>ASLWPAWFJZFCKF-UHFFFAOYSA-N | $3.8\times10^3$<br>$3.8\times10^3$<br>$4.1\times10^{-2}$<br>$1.3\times10^7$<br>3.0 | | HSDB (2015)<br>Zhang et al. (2010)<br>Zhang et al. (2010)<br>Zhang et al. (2010)<br>Zhang et al. (2010) | Q<br>Q<br>Q<br>Q<br>Q | 99<br>287, 288<br>287, 289<br>287, 290<br>287, 291 |
| tris(2-chloropropyl) phosphate<br>$C_9H_{18}Cl_3O_4P$<br>[6145-73-9]<br>GTRSAMFYSUBAGN-UHFFFAOYSA-N | $1.6\times10^2$<br>$1.4\times10^{-3}$<br>$6.7\times10^2$<br>$3.8\times10^{-1}$ | | Zhang et al. (2010)<br>Zhang et al. (2010)<br>Zhang et al. (2010)<br>Zhang et al. (2010) | Q<br>Q<br>Q<br>Q | 287, 288<br>287, 289<br>287, 290<br>287, 291 |
| tri-(2-chloroisopropyl)phosphate<br>$C_9H_{18}Cl_3O_4P$<br>[13674-84-5]<br>KVMPUXDNESXNOH-UHFFFAOYSA-N | $1.6\times10^2$<br>$1.6\times10^2$<br>$1.9\times10^{-4}$<br>$3.6\times10^4$<br>$3.8\times10^{-1}$ | | HSDB (2015)<br>Zhang et al. (2010)<br>Zhang et al. (2010)<br>Zhang et al. (2010)<br>Zhang et al. (2010) | Q<br>Q<br>Q<br>Q<br>Q | 99<br>287, 288<br>287, 289<br>287, 290<br>287, 291 |



Table A10.1: Phosphorus (C, H, O, N, Cl, Br, S, P) (...continued)

| Substance<br>Formula<br>(Trivial Name)<br>[CAS Registry Number]<br>InChIKey | $H_s^{cp}$<br>(at $T^\ominus$)<br>$\left[\dfrac{\text{mol}}{\text{m}^3\,\text{Pa}}\right]$ | $\dfrac{\text{d}\ln H_s^{cp}}{\text{d}(1/T)}$<br><br>[K] | Reference | Type | Note |
|---|---|---|---|---|---|
| bis(2-chloropropyl)<br>2-chloro-1-methylethyl phosphate<br>$C_9H_{18}Cl_3O_4P$<br>[76649-15-5]<br>WDLBXPUJJVVRGX-UHFFFAOYSA-N | $1.6\times10^2$<br><br>$7.2\times10^{-4}$<br>$5.6\times10^3$<br>$3.8\times10^{-1}$ | | Zhang et al. (2010)<br><br>Zhang et al. (2010)<br>Zhang et al. (2010)<br>Zhang et al. (2010) | Q<br><br>Q<br>Q<br>Q | 287, 288<br><br>287, 289<br>287, 290<br>287, 291 |
| tetrachlorvinphos<br>$C_{10}H_9Cl_4O_4P$<br>[22248-79-9]<br>UBCKGWBNUIFUST-YHYXMXQVSA-N | $5.5\times10^3$<br>$2.4\times10^3$<br>$5.4\times10^3$ | | HSDB (2015)<br>Ebert et al. (2023)<br>MacBean (2012a) | V<br>?<br>? | <br>316<br> |
| dimethylvinphos<br>$C_{10}H_{10}Cl_3O_4P$<br>[2274-67-1]<br>QSGNQELHULIMSJ-UHFFFAOYSA-N | $3.0\times10^2$ | | Ebert et al. (2023) | ? | 318 |
| diphenyl chlorophosphate<br>$C_{12}H_{10}ClO_3P$<br>[2524-64-3]<br>BHIIGRBMZRSDRI-UHFFFAOYSA-N | $3.7\times10^{-2}$ | | Bartelt-Hunt et al. (2008) | ? | 21 |
| chlorfenvinphos<br>$C_{12}H_{14}Cl_3O_4P$<br>(clofenvinfos)<br>[470-90-6]<br>FSAVDKDHPDSCTO-XYOKQWHBSA-N | $3.4\times10^2$<br>$3.4\times10^3$<br>$3.6\times10^3$<br>$3.5\times10^1$<br>4.1 | | HSDB (2015)<br>Mackay et al. (2006d)<br>Suntio et al. (1988)<br>Barcelo and Hennion (1997)<br>Goodarzi et al. (2010) | V<br>V<br>V<br>X<br>Q | <br><br>12<br>567<br>568 |
| crufomate<br>$C_{12}H_{19}ClNO_3P$<br>(ruelene)<br>[299-86-5]<br>BOFHKBLZOYVHSI-UHFFFAOYSA-N | $3.9\times10^3$ | | HSDB (2015) | Q | 99 |
| phosdiphen<br>$C_{14}H_{11}O_4Cl_4P$<br>[36519-00-3]<br>HEMINMLPKZELPP-UHFFFAOYSA-N | $2.6\times10^{-2}$ | | MacBean (2012a) | ? | |
| phosphamidon<br>$C_{10}H_{19}ClNO_5P$<br>[13171-21-6]<br>RGCLLPNLLBQHPF-HJWRWDBZSA-N | 2.8<br>2.8<br>$6.6\times10^6$ | | Mackay et al. (2006d)<br>Suntio et al. (1988)<br>HSDB (2015) | V<br>V<br>Q | <br>12<br>99 |
| tris(2,3-dibromo-1-propyl)<br>phosphate<br>$C_9H_{15}Br_6O_4P$<br>[126-72-7]<br>PQYJRMFWJJONBO-UHFFFAOYSA-N | $3.8\times10^{-1}$ | | HSDB (2015) | V | |



Table A10.1: Phosphorus (C, H, O, N, Cl, Br, S, P) (...continued)

| Substance Formula (Trivial Name) [CAS Registry Number] InChIKey | $H_s^{cp}$ (at $T^{\ominus}$) $\left[\dfrac{\text{mol}}{\text{m}^3\,\text{Pa}}\right]$ | $\dfrac{\text{d}\ln H_s^{cp}}{\text{d}(1/T)}$ [K] | Reference | Type | Note |
|---|---|---|---|---|---|
| naled C$_4$H$_7$Br$_2$Cl$_2$O$_4$P [300-76-5] BUYMVQAILCEWRR-UHFFFAOYSA-N | $1.5\times10^{-1}$ | | HSDB (2015) | V | |
| 2-bromo-1,1-dimethylethyl 2-bromoethyl 2-chloroethyl phosphate C$_9$H$_{18}$Br$_2$ClO$_4$P [125997-20-8] GZSKSYDWLZIPOX-UHFFFAOYSA-N | $1.5\times10^{3}$ $1.3\times10^{-2}$ $4.4\times10^{3}$ $8.2\times10^{-1}$ | | Zhang et al. (2010) Zhang et al. (2010) Zhang et al. (2010) Zhang et al. (2010) | Q Q Q Q | 287, 288 287, 289 287, 290 287, 291 |
| leptophos C$_{13}$H$_{10}$O$_3$BrCl$_2$P [21609-90-5] CVRALZAYCYJELZ-UHFFFAOYSA-N | 3.7 3.7 4.0 4.0 $2.6\times10^{1}$ | | Mackay and Shiu (1981) HSDB (2015) Mackay et al. (2006d) Suntio et al. (1988) Hilal et al. (2008) | L V V V Q | 12 |
| tributyl phosphorotrithioite C$_{12}$H$_{27}$PS$_3$ [150-50-5] KLAPGAOQRZTCBI-UHFFFAOYSA-N | $4.3\times10^{-1}$ $4.3\times10^{-1}$ $6.0\times10^{-4}$ $1.5\times10^{-1}$ $5.1\times10^{-2}$ | | HSDB (2015) Zhang et al. (2010) Zhang et al. (2010) Zhang et al. (2010) Zhang et al. (2010) | Q Q Q Q Q | 99 287, 288 287, 289 287, 290 287, 291 |
| bis(2,6,6-trimethylbicyclo[3.1.1]hept-2-enyl) bis(2,6,6-trimethylbicyclo[3.1.1]hept-2-enyl)thiodiphosphonate C$_{40}$H$_{60}$P$_2$S$_5$ [68400-79-3] ZZMOHCWFHCHISQ-UHFFFAOYSA-N | $8.2\times10^{-5}$ $5.8\times10^{4}$ $1.9\times10^{5}$ $1.6\times10^{2}$ | | Zhang et al. (2010) Zhang et al. (2010) Zhang et al. (2010) Zhang et al. (2010) | Q Q Q Q | 287, 288 287, 289 287, 290 287, 291 |
| thiometon C$_6$H$_{15}$O$_2$PS$_3$ [640-15-3] OPASCBHCTNRLRM-UHFFFAOYSA-N | $3.5\times10^{-1}$ | | HSDB (2015) | V | |
| demeton-S-methyl sulfone C$_6$H$_{15}$O$_5$PS$_2$ [17040-19-6] PZIRJMYRYORVIT-UHFFFAOYSA-N | $<2.3\times10^{10}$ | | MacBean (2012a) | ? | |
| oxydemeton-methyl C$_6$H$_{15}$O$_4$PS$_2$ [301-12-2] PMCVMORKVPSKHZ-UHFFFAOYSA-N | $6.2\times10^{7}$ | | HSDB (2015) | Q | 99 |



Table A10.1: Phosphorus (C, H, O, N, Cl, Br, S, P) (...continued)

| Substance<br>Formula<br>(Trivial Name)<br>[CAS Registry Number]<br>InChIKey | $H_s^{cp}$<br>(at $T^\ominus$)<br>$\left[\dfrac{\text{mol}}{\text{m}^3\,\text{Pa}}\right]$ | $\dfrac{\text{d}\ln H_s^{cp}}{\text{d}(1/T)}$<br><br>[K] | Reference | Type | Note |
|---|---|---|---|---|---|
| demeton-O-methyl<br>$C_6H_{15}O_3PS_2$<br>[867-27-6]<br>ZVZQKNVMDKSGGF-UHFFFAOYSA-N | 3.2 | | Ebert et al. (2023) | ? | 316 |
| demeton-S-methyl<br>$C_6H_{15}O_3PS_2$<br>[919-86-8]<br>WEBQKRLKWNIYKK-UHFFFAOYSA-N | $3.7\times10^2$<br>$3.6\times10^2$ | | HSDB (2015)<br>Mackay et al. (2006d) | V<br>V | |
| methacrifos<br>$C_7H_{13}O_5PS$<br>[62610-77-9]<br>NTAHCMPOMKHKEU-AATRIKPKSA-N | $1.0\times10^1$ | | MacBean (2012a) | ? | |
| phorate<br>$C_7H_{17}O_2PS_3$<br>[298-02-2]<br>BULVZWIRKLYCBC-UHFFFAOYSA-N | 2.1<br>$9.9\times10^{-1}$<br>1.5<br>$1.5\times10^{-2}$<br>$4.5\times10^{-2}$ | | HSDB (2015)<br>Mackay et al. (2006d)<br>Suntio et al. (1988)<br>Barcelo and Hennion (1997)<br>Goodarzi et al. (2010) | V<br>V<br>V<br>X<br>Q | <br><br>12<br>567<br>568 |
| salithion<br>$C_8H_9O_3PS$<br>[3811-49-2]<br>OUNSASXJZHBGAI-UHFFFAOYSA-N | $4.7\times10^{-1}$ | | MacBean (2012a) | ? | |
| acetoxon<br>$C_8H_{17}O_5PS$<br>[2425-25-4]<br>ZRCQYAQOWIQUBA-UHFFFAOYSA-N | $1.3\times10^4$ | | HSDB (2015) | Q | 99 |
| demeton-O<br>$C_8H_{19}O_3PS_2$<br>[298-03-3]<br>DGLIBALSRMUQDD-UHFFFAOYSA-N | 6.1 | | MacBean (2012a) | ? | 12 |
| demeton-S<br>$C_8H_{19}O_3PS_2$<br>(isosystox)<br>[126-75-0]<br>GRPRVIYRYGLIJU-UHFFFAOYSA-N | $2.0\times10^2$ | | HSDB (2015) | V | |
| sulfotep<br>$C_8H_{20}O_5P_2S_2$<br>[3689-24-5]<br>XIUROWKZWPIAIB-UHFFFAOYSA-N | 2.2<br>3.4<br>$9.9\times10^{-3}$<br>$4.4\times10^{-3}$ | | HSDB (2015)<br>Mackay et al. (2006d)<br>Barcelo and Hennion (1997)<br>Goodarzi et al. (2010) | V<br>V<br>X<br>Q | <br><br>567<br>568, 571 |





Table A10.1: Phosphorus (C, H, O, N, Cl, Br, S, P) (... continued)

| Substance<br>Formula<br>(Trivial Name)<br>[CAS Registry Number]<br>InChIKey | $H_s^{cp}$<br>(at $T^{\ominus}$)<br>$\left[\dfrac{\text{mol}}{\text{m}^3\,\text{Pa}}\right]$ | $\dfrac{\text{d}\ln H_s^{cp}}{\text{d}(1/T)}$<br><br>[K] | Reference | Type | Note |
|---|---|---|---|---|---|
| tetrakis(hydroxymethyl)<br>phosphonium sulfate<br>$C_8H_{24}O_{12}P_2S$<br>[55566-30-8]<br>YIEDHPBKGZGLIK-UHFFFAOYSA-L | $5.8\times10^{17}$ | | HSDB (2015) | Q | 99 |
| ethoprophos<br>$C_8H_{19}O_2PS_2$<br>[13194-48-4]<br>VJYFKVYYMZPMAB-UHFFFAOYSA-N | $6.1\times10^1$<br>$6.1\times10^1$ | | HSDB (2015)<br>Mackay et al. (2006d) | V<br>V | |
| disulfoton<br>$C_8H_{19}O_2PS_3$<br>[298-04-4]<br>DOFZAZXDOSGAJZ-UHFFFAOYSA-N | $1.1\times10^1$<br>4.5<br>4.5<br>4.5<br>$4.5\times10^{-2}$<br>$6.4\times10^{-1}$ | | Muir et al. (2004)<br>HSDB (2015)<br>Mackay et al. (2006d)<br>Suntio et al. (1988)<br>Barcelo and Hennion (1997)<br>Goodarzi et al. (2010) | L<br>V<br>V<br>V<br>X<br>Q | 367<br><br><br>12<br>567<br>568, 571 |
| endothion<br>$C_9H_{13}O_6PS$<br>[2778-04-3]<br>YCAGGFXSFQFVQL-UHFFFAOYSA-N | $1.5\times10^6$ | | HSDB (2015) | Q | 99 |
| terbufos<br>$C_9H_{21}O_2PS_3$<br>[13071-79-9]<br>XLNZEKHULJKQBA-UHFFFAOYSA-N | $4.1\times10^{-1}$<br>$4.1\times10^{-1}$<br>$9.9\times10^{-3}$<br>$3.0\times10^{-1}$ | | HSDB (2015)<br>Mackay et al. (2006d)<br>Barcelo and Hennion (1997)<br>Goodarzi et al. (2010) | V<br>V<br>X<br>Q | <br><br>567<br>568 |
| ethion<br>$C_9H_{22}O_4P_2S_4$<br>[563-12-2]<br>RIZMRRKBZQXFOY-UHFFFAOYSA-N | $2.6\times10^1$<br>$3.1\times10^1$<br>$3.1\times10^1$<br>$3.1\times10^{-1}$<br>$6.8\times10^{-1}$ | | HSDB (2015)<br>Mackay et al. (2006d)<br>Suntio et al. (1988)<br>Barcelo and Hennion (1997)<br>Goodarzi et al. (2010) | V<br>V<br>V<br>X<br>Q | <br><br>12<br>567<br>568 |
| fonofos<br>$C_{10}H_{15}OPS_2$<br>[944-22-9]<br>KVGLBTYUCJYMND-UHFFFAOYSA-N | 1.4<br>1.4<br>1.4<br>$1.9\times10^5$ | | Duchowicz et al. (2020)<br>HSDB (2015)<br>Mackay et al. (2006d)<br>Duchowicz et al. (2020) | V<br>V<br>V<br>Q | 186 |
| fenthion<br>$C_{10}H_{15}O_3PS_2$<br>[55-38-9]<br>PNVJTZOFSHSLTO-UHFFFAOYSA-N | 6.8<br>$4.5\times10^1$<br>$4.5\times10^1$<br>$4.5\times10^{-1}$<br>1.9 | | HSDB (2015)<br>Mackay et al. (2006d)<br>Suntio et al. (1988)<br>Barcelo and Hennion (1997)<br>Goodarzi et al. (2010) | V<br>V<br>V<br>X<br>Q | <br><br>12<br>567<br>568 |



Table A10.1: Phosphorus (C, H, O, N, Cl, Br, S, P) (...continued)

| Substance<br>Formula<br>(Trivial Name)<br>[CAS Registry Number]<br>InChIKey | $H_s^{cp}$<br>(at $T^{\ominus}$)<br>$\left[\dfrac{\text{mol}}{\text{m}^3\,\text{Pa}}\right]$ | $\dfrac{\text{d}\ln H_s^{cp}}{\text{d}(1/T)}$<br><br>[K] | Reference | Type | Note |
|---|---|---|---|---|---|
| malathion<br>$C_{10}H_{19}O_6PS_2$<br>[121-75-5]<br>JXSJBGJIGXNWCI-UHFFFAOYSA-N | $2.0\times10^1$<br>$1.5\times10^1$<br>$6.7\times10^2$<br>$2.0\times10^3$<br>$4.4\times10^2$<br>$2.5\times10^2$<br>$4.3\times10^2$<br>$1.7\times10^2$<br>$7.3\times10^3$<br>$2.6\times10^1$<br>$4.3$<br>$1.8$<br>$1.5\times10^2$<br>$2.0\times10^3$ | 8300 | Brockbank (2013)<br>Chao et al. (2017)<br>Watanabe (1993)<br>Fendinger and Glotfelty (1990)<br>Mackay et al. (2006d)<br>Cotham and Bidleman (1989)<br>Suntio et al. (1988)<br>Glotfelty et al. (1987)<br>Sanders and Seiber (1983)<br>Mackay and Shiu (1981)<br>Barcelo and Hennion (1997)<br>Goodarzi et al. (2010)<br>Hilal et al. (2008)<br>Bartelt-Hunt et al. (2008) | L<br>M<br>M<br>M<br>V<br>V<br>V<br>V<br>V<br>V<br>X<br>Q<br>Q<br>? | 1<br><br><br><br><br><br>12<br><br>87<br><br>567<br>568<br><br>21 |
| malaoxon<br>$C_{10}H_{19}O_7PS$<br>[1634-78-2]<br>WSORODGWGUUOBO-UHFFFAOYSA-N | $5.5\times10^6$ | | HSDB (2015) | Q | 99 |
| cadusafos<br>$C_{10}H_{23}O_2PS_2$<br>[95465-99-9]<br>KXRPCFINVWWFHQ-UHFFFAOYSA-N | $7.6$ | | HSDB (2015) | V | |
| fensulfothion<br>$C_{11}H_{17}O_4PS_2$<br>[115-90-2]<br>XDNBJTQLKCIJBV-UHFFFAOYSA-N | $7.0\times10^4$ | | HSDB (2015) | Q | 99 |
| phenthoate<br>$C_{12}H_{17}O_4PS_2$<br>[2597-03-7]<br>XAMUDJHXFNRLCY-UHFFFAOYSA-N | $1.8\times10^3$<br>$9.8\times10^1$ | | HSDB (2015)<br>Mackay et al. (2006d) | V<br>V | |
| sulprofos<br>$C_{12}H_{19}O_2PS_3$<br>[35400-43-2]<br>JXHJNEJVUNHLKO-UHFFFAOYSA-N | $1.1\times10^1$<br>$1.1\times10^1$ | | HSDB (2015)<br>MacBean (2012a) | V<br>? | <br>12 |
| S,S,S-tributyl phosphorotrithioate<br>$C_{12}H_{27}OPS_3$<br>(DEF)<br>[78-48-8]<br>ZOKXUAHZSKEQSS-UHFFFAOYSA-N | $3.4\times10^1$<br>$1.3$ | | Fendinger and Glotfelty (1990)<br>Glotfelty et al. (1987) | M<br>V | |



Table A10.1: Phosphorus (C, H, O, N, Cl, Br, S, P) (. . . continued)

| Substance<br>Formula<br>(Trivial Name)<br>[CAS Registry Number]<br>InChIKey | $H_s^{cp}$<br>(at $T^{\ominus}$)<br><br>$\left[\dfrac{\mathrm{mol}}{\mathrm{m^3\,Pa}}\right]$ | $\dfrac{\mathrm{d}\ln H_s^{cp}}{\mathrm{d}(1/T)}$<br><br>[K] | Reference | Type | Note |
|---|---|---|---|---|---|
| aspon<br>$C_{12}H_{28}O_5P_2S_2$<br>(tetrapropyl dithiopyrophosphate)<br>[3244-90-4]<br>IIDFEIDMIKSJSV-UHFFFAOYSA-N | 6.1 | | Ebert et al. (2023) | ? | 318 |
| iprobenphos<br>$C_{13}H_{21}O_3PS$<br>[26087-47-8]<br>FCOAHACKGGIURQ-UHFFFAOYSA-N | $2.6\times10^2$ | | Watanabe (1993) | M | |
| propaphos<br>$C_{13}H_{21}O_4PS$<br>[7292-16-2]<br>PWYIUEFFPNVCMW-UHFFFAOYSA-N | $3.4\times10^3$<br>$3.4\times10^3$<br>$3.6\times10^1$<br>$3.4\times10^3$ | | Duchowicz et al. (2020)<br>HSDB (2015)<br>Duchowicz et al. (2020)<br>MacBean (2012a) | V<br>V<br>Q<br>? | 186 |
| edifenphos<br>$C_{14}H_{15}O_2PS_2$<br>[17109-49-8]<br>AWZOLILCOUMRDG-UHFFFAOYSA-N | $5.0\times10^3$<br>$1.3\times10^4$<br>$1.4\times10^1$ | | Watanabe (1993)<br>HSDB (2015)<br>Mackay et al. (2006d) | M<br>V<br>V | |
| systox<br>$C_{16}H_{38}O_6P_2S_4$<br>[8065-48-3]<br>FAXIJTUDSBIMHY-UHFFFAOYSA-N | $5.5\times10^1$ | | HSDB (2015) | V | |
| temefos<br>$C_{16}H_{20}O_6P_2S_3$<br>(temephos)<br>[3383-96-8]<br>WWJZWCUNLNYYAU-UHFFFAOYSA-N | 1.7<br>$4.9\times10^3$<br>4.4 | | Barcelo and Hennion (1997)<br>HSDB (2015)<br>Goodarzi et al. (2010) | X<br>Q<br>Q | 567<br>99<br>568 |
| methamidophos<br>$C_2H_8NOPS_2$<br>[10265-92-6]<br>NNKVPIKMPCQWCG-UHFFFAOYSA-N | $1.1\times10^4$ | | HSDB (2015) | Q | 99 |
| acephate<br>$C_4H_{10}NO_3PS$<br>[30560-19-1]<br>YASYVMFAVPKPKE-UHFFFAOYSA-N | $2.0\times10^7$<br>$2.0\times10^7$<br>$1.9\times10^5$<br>$4.4\times10^2$ | | HSDB (2015)<br>Mackay et al. (2006d)<br>Barcelo and Hennion (1997)<br>Goodarzi et al. (2010) | V<br>V<br>X<br>Q | 567<br>568 |
| dimethoate<br>$C_5H_{12}NO_3PS_2$<br>[60-51-5]<br>MCWXGJITAZMZEV-UHFFFAOYSA-N | $4.1\times10^4$<br>$8.7\times10^3$<br>$9.1\times10^3$<br>$9.0\times10^1$<br>$5.3\times10^2$ | | HSDB (2015)<br>Mackay et al. (2006d)<br>Suntio et al. (1988)<br>Barcelo and Hennion (1997)<br>Goodarzi et al. (2010) | V<br>V<br>V<br>X<br>Q | 12<br>567<br>568, 569 |





Table A10.1: Phosphorus (C, H, O, N, Cl, Br, S, P) (. . . continued)

| Substance / Formula / (Trivial Name) / [CAS Registry Number] / InChIKey | $H_s^{cp}$ (at $T^\ominus$) $\left[\dfrac{\mathrm{mol}}{\mathrm{m^3\,Pa}}\right]$ | $\dfrac{\mathrm{d}\ln H_s^{cp}}{\mathrm{d}(1/T)}$ [K] | Reference | Type | Note |
|---|---|---|---|---|---|
| omethoate $C_5H_{12}NO_4PS$ [1113-02-6] PZXOQEXFMJCDPG-UHFFFAOYSA-N | $2.1\times10^8$ | | HSDB (2015) | Q | 99 |
| methidathion $C_6H_{11}N_2O_4PS_3$ [950-37-8] MEBQXILRKZHVCX-UHFFFAOYSA-N | $1.4\times10^3$ $5.8\times10^3$ $5.8\times10^3$ | | HSDB (2015) Glotfelty et al. (1987) Burkhard and Guth (1981) | V V V | |
| fosthietan $C_6H_{12}NO_3PS_2$ [21548-32-3] RHJOIOVESMTJEK-UHFFFAOYSA-N | $2.4\times10^5$ $2.4\times10^5$ | | HSDB (2015) MacBean (2012a) | V ? | |
| formothion $C_6H_{12}NO_4PS_2$ [2540-82-1] AIKKULXCBHRFOS-UHFFFAOYSA-N | $9.0\times10^4$ | | HSDB (2015) | V | |
| menazon $C_6H_{12}N_5O_2PS_2$ [78-57-9] SUYHYHLFUHHVJQ-UHFFFAOYSA-N | $6.6\times10^3$ | | HSDB (2015) | V | |
| ethoate-methyl $C_6H_{14}NO_3PS_2$ [116-01-8] DICRHEJCQXFJBY-UHFFFAOYSA-N | $3.5\times10^5$ | | HSDB (2015) | Q | 99 |
| glyphosate-trimesium $C_6H_{16}NO_5PS$ [81591-81-3] RUCAXVJJQQJZGU-UHFFFAOYSA-M | $>2.3\times10^{10}$ | | MacBean (2012a) | ? | |
| phosfolan $C_7H_{14}NO_3PS_2$ (cyolane) [947-02-4] ILBONRFSLATCRE-UHFFFAOYSA-N | $5.1\times10^4$ | | Ebert et al. (2023) | ? | 316 |
| methylparathion $C_8H_{10}NO_5PS$ (parathion-methyl) [298-00-0] RLBIQVVOMOPOHC-UHFFFAOYSA-N | $5.0\times10^1$ $1.7\times10^1$ $2.6\times10^2$ $1.6\times10^2$ $9.9\times10^1$ $4.7\times10^1$ $9.9\times10^1$ $4.7\times10^1$ $9.2\times10^1$ $2.1\times10^3$ | | Mackay and Shiu (1981) Chao et al. (2017) Rice et al. (1997b) Fendinger and Glotfelty (1990) Metcalfe et al. (1980) Mackay et al. (2006d) Woodrow et al. (1990) Suntio et al. (1988) Glotfelty et al. (1987) Sanders and Seiber (1983) | L M M M M V V V V V | 12 12 87 |



Table A10.1: Phosphorus (C, H, O, N, Cl, Br, S, P) (...continued)

| Substance Formula (Trivial Name) [CAS Registry Number] InChIKey | $H_s^{cp}$ (at $T^\ominus$) $\left[\dfrac{\mathrm{mol}}{\mathrm{m^3\,Pa}}\right]$ | $\dfrac{\mathrm{d}\ln H_s^{cp}}{\mathrm{d}(1/T)}$ [K] | Reference | Type | Note |
|---|---|---|---|---|---|
| | $1.6\times10^2$ | | Metcalfe et al. (1980) | V | |
| | $1.3\times10^{-1}$ | | Smith and Bomberger (1980) | V | 24 |
| | $4.7\times10^{-1}$ | | Barcelo and Hennion (1997) | X | 567 |
| | $2.1\times10^1$ | | Goodarzi et al. (2010) | Q | 568 |
| | $1.5\times10^1$ | | Hilal et al. (2008) | Q | |
| zinophos $C_8H_{13}N_2O_3PS$ (thionazin) [297-97-2] IRVDMKJLOCGUBJ-UHFFFAOYSA-N | $1.0\times10^1$ $1.2\times10^1$ $1.1\times10^1$ | | Mackay et al. (2006d) Suntio et al. (1988) MacBean (2012a) | V V ? | 12 |
| vamidothion $C_8H_{18}NO_4PS_2$ [2275-23-2] LESVOLZBIFDZGS-UHFFFAOYSA-N | $1.1\times10^{10}$ | | HSDB (2015) | Q | 99 |
| cyanophos $C_9H_{10}NO_3PS$ [2636-26-2] SCKHCCSZFPSHGR-UHFFFAOYSA-N | $1.8$ | | HSDB (2015) | V | |
| fenitrothion $C_9H_{12}NO_5PS$ [122-14-5] ZNOLGFHPUIJIMJ-UHFFFAOYSA-N | $8.3\times10^1$ $1.1\times10^1$ $8.3\times10^2$ $2.8\times10^2$ $2.7\times10^1$ $1.5\times10^1$ $2.7$ $4.4\times10^1$ $5.3$ | | Watanabe (1993) Metcalfe et al. (1980) Mackay et al. (2006d) Suntio et al. (1988) Mackay and Shiu (1981) Metcalfe et al. (1980) Barcelo and Hennion (1997) Goodarzi et al. (2010) Hilal et al. (2008) | M M V V V V X Q Q | 12 567 568, 569 |
| fosthiazate-1 $C_9H_{18}NO_3PS_2$ [98886-44-3] DUFVKSUJRWYZQP-UHFFFAOYSA-N | $7.5\times10^1$ $1.0\times10^5$ $5.7\times10^4$ $5.7\times10^4$ $5.7\times10^4$ | | MacBean (2012b) Keshavarz et al. (2022) Duchowicz et al. (2020) Duchowicz et al. (2020) Maniere et al. (2011) | X Q Q ? ? | 350 185, 21 241, 165 |
| prothoate $C_9H_{20}NO_3PS_2$ (trimethoate) [2275-18-5] QTXHFDHVLBDJIO-UHFFFAOYSA-N | $1.5\times10^5$ | | HSDB (2015) | Q | 99 |
| amiton $C_{10}H_{24}NO_3PS$ [78-53-5] PJISLFCKHOHLLP-UHFFFAOYSA-N | $3.5\times10^4$ | | Bartelt-Hunt et al. (2008) | ? | 21 |



Table A10.1: Phosphorus (C, H, O, N, Cl, Br, S, P) (...continued)

| Substance<br>Formula<br>(Trivial Name)<br>[CAS Registry Number]<br>InChIKey | $H_s^{cp}$<br>(at $T^{\ominus}$)<br>$\left[\dfrac{\text{mol}}{\text{m}^3\,\text{Pa}}\right]$ | $\dfrac{\mathrm{d}\ln H_s^{cp}}{\mathrm{d}(1/T)}$<br><br>[K] | Reference | Type | Note |
|---|---|---|---|---|---|
| azinphos-methyl<br>$C_{10}H_{12}N_3O_3PS_2$<br>[86-50-0]<br>CJJOSEISRRTUQB-UHFFFAOYSA-N | $3.4\times10^3$<br>$3.2\times10^3$<br>$3.1\times10^2$<br>$3.1$<br>$4.6\times10^1$ | | HSDB (2015)<br>Mackay et al. (2006d)<br>Suntio et al. (1988)<br>Barcelo and Hennion (1997)<br>Goodarzi et al. (2010) | V<br>V<br>V<br>X<br>Q | <br><br>12<br>567<br>568 |
| parathion<br>$C_{10}H_{14}NO_5PS$<br>(E 605)<br>[56-38-2]<br>LCCNCVORNKJIRZ-UHFFFAOYSA-N | $1.2\times10^2$<br>$7.1\times10^1$<br>$5.0\times10^1$<br>$5.0\times10^1$<br>$8.3\times10^1$<br>$4.2\times10^1$<br>$1.6\times10^3$<br>$8.1$<br>$1.3\times10^1$<br>$1.0\times10^1$<br>$3.3\times10^1$<br>$8.0\times10^{-1}$<br>$3.2\times10^{-1}$<br>$6.5$<br>$3.4\times10^1$ | | Fendinger and Glotfelty (1990)<br>Mackay et al. (2006d)<br>Siebers and Mattusch (1996)<br>Siebers et al. (1994)<br>Suntio et al. (1988)<br>Glotfelty et al. (1987)<br>Sanders and Seiber (1983)<br>Mackay and Shiu (1981)<br>Burkhard and Guth (1981)<br>Chiou et al. (1980)<br>MacBean (2012b)<br>Barcelo and Hennion (1997)<br>Goodarzi et al. (2010)<br>Hilal et al. (2008)<br>Bartelt-Hunt et al. (2008) | M<br>V<br>V<br>V<br>V<br>V<br>V<br>V<br>V<br>V<br>X<br>X<br>Q<br>Q<br>? | <br><br>12<br><br>12<br><br>87<br><br><br><br>350<br>567<br>568<br><br>21 |
| isoparathion<br>$C_{10}H_{14}NO_5PS$<br>[597-88-6]<br>BGWJTLLALYACOG-UHFFFAOYSA-N | $6.1\times10^1$ | | Ebert et al. (2023) | ? | 318 |
| etrimfos<br>$C_{10}H_{17}N_2O_4PS$<br>[38260-54-7]<br>FGIWFCGDPUIBEZ-UHFFFAOYSA-N | $1.6\times10^1$ | | HSDB (2015) | V | |
| propetamphos<br>$C_{10}H_{20}NO_4PS$<br>[31218-83-4]<br>BZNDWPRGXNILMS-VQHVLOKHSA-N | $2.1\times10^2$ | | HSDB (2015) | V | |
| mecarbam<br>$C_{10}H_{20}NO_5PS_2$<br>[2595-54-2]<br>KLGMSAOQDHLCOS-UHFFFAOYSA-N | $1.1\times10^4$ | | HSDB (2015) | Q | 99 |
| phosmet<br>$C_{11}H_{12}NO_4PS_2$<br>[732-11-6]<br>LMNZTLDVJIUSHT-UHFFFAOYSA-N | $1.2\times10^3$<br>$1.3\times10^3$<br>$1.1\times10^3$<br>$1.0\times10^1$<br>$3.8$<br>$7.4\times10^2$ | | HSDB (2015)<br>Mackay et al. (2006d)<br>Suntio et al. (1988)<br>Barcelo and Hennion (1997)<br>Goodarzi et al. (2010)<br>Maniere et al. (2011) | V<br>V<br>V<br>X<br>Q<br>? | <br><br>12<br>567<br>568<br>165 |



Table A10.1: Phosphorus (C, H, O, N, Cl, Br, S, P) (. . . continued)

| Substance Formula (Trivial Name) [CAS Registry Number] InChIKey | $H_s^{cp}$ (at $T^{\ominus}$) $\left[\dfrac{\mathrm{mol}}{\mathrm{m^3\,Pa}}\right]$ | $\dfrac{\mathrm{d}\ln H_s^{cp}}{\mathrm{d}(1/T)}$ [K] | Reference | Type | Note |
|---|---|---|---|---|---|
| pirimiphos methyl $C_{11}H_{20}N_3O_3PS$ [29232-93-7] QHOQHJPRIBSPCY-UHFFFAOYSA-N | $1.6\times10^1$ $1.7\times10^1$ | | HSDB (2015) Maniere et al. (2011) | V ? | 241, 165 |
| imicyafos $C_{11}H_{21}N_4O_2PS$ [140163-89-9] PPCUNNLZTNMXFO-UHFFFAOYSA-N | $1.5\times10^9$ | | Ebert et al. (2023) | ? | 318 |
| Agent VX $C_{11}H_{26}NO_2PS$ [50782-69-9] JJIUCEJQJXNMHV-UHFFFAOYSA-N | $9.1\times10^2$ $2.8\times10^3$ $1.2\times10^3$ | | HSDB (2015) Bartelt-Hunt et al. (2008) Opresko et al. (1998) | V ? ? | 21 |
| ditalimfos $C_{12}H_{14}NO_4PS$ [5131-24-8] MTBZIGHNGSTDJV-UHFFFAOYSA-N | $2.3\times10^3$ | | Ebert et al. (2023) | ? | 316 |
| quinalphos $C_{12}H_{15}N_2O_3PS$ [13593-03-8] JYQUHIFYBATCCY-UHFFFAOYSA-N | $1.1\times10^2$ | | Ebert et al. (2023) | ? | 316 |
| phoxim $C_{12}H_{15}N_2O_3PS$ [14816-18-3] ATROHALUCMTWTB-UHFFFAOYSA-N | $5.0\times10^1$ | | Ebert et al. (2023) | ? | 316 |
| triazophos $C_{12}H_{16}N_3O_3PS$ [24017-47-8] AMFGTOFWMRQMEM-UHFFFAOYSA-N | $3.2\times10^2$ | | HSDB (2015) | V | |
| azinphos-ethyl $C_{12}H_{16}N_3O_3PS_2$ [2642-71-9] RQVGAIADHNPSME-UHFFFAOYSA-N | $1.0\times10^2$ | | HSDB (2015) | V | |
| diazinon $C_{12}H_{21}N_2O_3PS$ (dimpylate) [333-41-5] FHIVAFMUCKRCQO-UHFFFAOYSA-N | $4.6\times10^1$ $9.2\times10^1$ $1.5\times10^1$ $1.1\times10^1$ $8.4\times10^1$ $8.8\times10^1$ $2.5\times10^1$ $1.5\times10^1$ $6.7$ $1.0\times10^2$ $1.3\times10^1$ | 12000 | Muir et al. (2004) Muir et al. (2004) Feigenbrugel et al. (2004a) Watanabe (1993) Fendinger et al. (1989) Fendinger and Glotfelty (1988) Mackay et al. (2006d) Suntio et al. (1988) Glotfelty et al. (1987) Sanders and Seiber (1983) Burkhard and Guth (1981) | L L M M M M V V V V V | 367 366 72 72 12 87 |





Table A10.1: Phosphorus (C, H, O, N, Cl, Br, S, P) (...continued)

| Substance Formula (Trivial Name) [CAS Registry Number] InChIKey | $H_s^{cp}$ (at $T^\ominus$) $\left[\dfrac{\mathrm{mol}}{\mathrm{m}^3\,\mathrm{Pa}}\right]$ | $\dfrac{\mathrm{d}\ln H_s^{cp}}{\mathrm{d}(1/T)}$ [K] | Reference | Type | Note |
|---|---|---|---|---|---|
| | $1.5\times10^{-1}$ | | Barcelo and Hennion (1997) | X | 567 |
| | $1.7\times10^{-1}$ | | Goodarzi et al. (2010) | Q | 568 |
| | $1.4\times10^{2}$ | | Meylan and Howard (1991) | Q | |
| isoxathion C$_{13}$H$_{16}$NO$_4$PS [18854-01-8] SDMSCIWHRZJSRN-UHFFFAOYSA-N | $1.6\times10^{2}$ | | HSDB (2015) | Q | 99 |
| butamifos C$_{13}$H$_{21}$N$_2$O$_4$PS [36335-67-8] OEYOMNZEMCPTKN-UHFFFAOYSA-N | $3.4\times10^{1}$ | | Ebert et al. (2023) | ? | 316 |
| fenamiphos C$_{13}$H$_{22}$NO$_3$PS [22224-92-6] ZCJPOPBZHLUFHF-UHFFFAOYSA-N | $1.1\times10^{3}$ | | HSDB (2015) | V | |
| tebupirimfos C$_{13}$H$_{23}$N$_2$O$_3$PS [96182-53-5] AWYOMXWDGWUJHS-UHFFFAOYSA-N | 3.5 | | HSDB (2015) | V | |
| pirimiphos ethyl C$_{13}$H$_{24}$N$_3$O$_3$PS [23505-41-1] TZBPRYIIJAJUOY-UHFFFAOYSA-N | $1.8\times10^{-1}$ | | HSDB (2015) | V | |
| bensulide C$_{14}$H$_{24}$NO$_4$PS$_3$ [741-58-2] RRNIZKPFKNDSRS-UHFFFAOYSA-N | $1.1\times10^{3}$ | | HSDB (2015) | V | |
| ethyl $p$-nitrophenyl benzenethiophosphonate C$_{14}$H$_{14}$NO$_4$PS [2104-64-5] AIGRXSNSLVJMEA-UHFFFAOYSA-N | $2.2\times10^{1}$ $2.2\times10^{1}$ $4.3\times10^{7}$ | | Duchowicz et al. (2020) HSDB (2015) Duchowicz et al. (2020) | V V Q | 186 |
| pyridaphenthion C$_{14}$H$_{17}$N$_2$O$_4$PS [119-12-0] CXJSOEPQXUCJSA-UHFFFAOYSA-N | $5.1\times10^{4}$ | | Ebert et al. (2023) | ? | 318 |
| piperophos C$_{14}$H$_{28}$NO$_3$PS$_2$ [24151-93-7] UNLYSVIDNRIVFJ-UHFFFAOYSA-N | $8.8\times10^{2}$ | | Ebert et al. (2023) | ? | 318 |





Table A10.1: Phosphorus (C, H, O, N, Cl, Br, S, P) (...continued)

| Substance Formula (Trivial Name) [CAS Registry Number] InChIKey | $H_s^{cp}$ (at $T^{\ominus}$) $\left[\dfrac{\mathrm{mol}}{\mathrm{m^3\,Pa}}\right]$ | $\dfrac{\mathrm{d}\ln H_s^{cp}}{\mathrm{d}(1/T)}$ [K] | Reference | Type | Note |
|---|---|---|---|---|---|
| isofenphos $C_{15}H_{24}NO_4PS$ [25311-71-1] HOQADATXFBOEGG-UHFFFAOYSA-N | $1.2\times10^2$ $2.4\times10^2$ $1.0\times10^3$ $6.9\times10^{-1}$ $2.4\times10^2$ | | Mackay et al. (2006d) MacBean (2012b) Barcelo and Hennion (1997) Goodarzi et al. (2010) MacBean (2012a) | V X X Q ? | 350 567 568, 569 12 |
| chlormephos $C_5H_{12}ClO_2PS_2$ [24934-91-6] QGTYWWGEWOBMAK-UHFFFAOYSA-N | $3.4\times10^{-2}$ $3.4\times10^{-2}$ $3.2\times10^5$ | | Duchowicz et al. (2020) HSDB (2015) Duchowicz et al. (2020) | V V Q | 186 |
| chlorethoxyfos $C_6H_{11}Cl_4O_3PS$ [54593-83-8] XFDJMIHUAHSGKG-UHFFFAOYSA-N | $2.3$ | | HSDB (2015) | Q | 99 |
| ronnel $C_8H_8O_3Cl_3PS$ [299-84-3] JHJOOSLFWRRSGU-UHFFFAOYSA-N | $4.8\times10^{-1}$ $1.7\times10^{-2}$ $3.1\times10^{-1}$ $5.7\times10^{-2}$ | | Mackay and Shiu (1981) Mackay et al. (2006d) Suntio et al. (1988) Hilal et al. (2008) | L V V Q | 12 |
| tolclofos-methyl $C_9H_{11}Cl_2O_3PS$ [57018-04-9] OBZIQQJJIKNWNO-UHFFFAOYSA-N | $1.7\times10^{-2}$ $2.7$ | | Mackay et al. (2006d) Maniere et al. (2011) | V ? | 12, 165 |
| methyl trithion $C_9H_{12}ClO_2PS_3$ [953-17-3] OUCCVXVYGFBXSV-UHFFFAOYSA-N | $9.9\times10^1$ | | HSDB (2015) | Q | 99 |
| trichloronate $C_{10}H_{12}Cl_3O_2PS$ [327-98-0] ANIAQSUBRGXWLS-UHFFFAOYSA-N | $8.8\times10^{-1}$ $9.0\times10^{-1}$ $2.6\times10^4$ $7.5\times10^1$ | | Duchowicz et al. (2020) HSDB (2015) Duchowicz et al. (2020) MacBean (2012a) | V V Q ? | 186 |
| dichlofenthion $C_{10}H_{13}Cl_2O_3PS$ [97-17-6] WGOWCPGHOCIHBW-UHFFFAOYSA-N | $1.0\times10^{-2}$ $1.0\times10^{-2}$ $3.2\times10^{-5}$ $3.2\times10^{-5}$ $1.5\times10^1$ | | Duchowicz et al. (2020) HSDB (2015) Mackay et al. (2006d) Suntio et al. (1988) Duchowicz et al. (2020) | V V V V Q | 186 12 |
| prothiofos $C_{11}H_{15}Cl_2O_2PS_2$ [34643-46-4] FITIWKDOCAUBQD-UHFFFAOYSA-N | $3.4\times10^{-1}$ | | Ebert et al. (2023) | ? | 318 |
| chlorthiophos $C_{11}H_{15}Cl_2O_3PS_2$ [21923-23-9] JAZJVWLGNLCNDD-UHFFFAOYSA-N | $8.2$ | | HSDB (2015) | Q | 99 |



Table A10.1: Phosphorus (C, H, O, N, Cl, Br, S, P) (...continued)

| Substance / Formula / (Trivial Name) / [CAS Registry Number] / InChIKey | $H_s^{cp}$ (at $T^{\ominus}$) $\left[\dfrac{\mathrm{mol}}{\mathrm{m^3\,Pa}}\right]$ | $\dfrac{\mathrm{d}\ln H_s^{cp}}{\mathrm{d}(1/T)}$ [K] | Reference | Type | Note |
|---|---|---|---|---|---|
| carbophenothion $C_{11}H_{16}ClO_2PS_3$ [786-19-6] VEDTXTNSFWUXGQ-UHFFFAOYSA-N | $4.9\times10^1$ $2.2\times10^1$ | | HSDB (2015) Suntio et al. (1988) | V V | 12 |
| coumaphos $C_{14}H_{16}ClO_5PS$ [56-72-4] BXNANOICGRISHX-UHFFFAOYSA-N | $9.0\times10^1$ | | HSDB (2015) | V | |
| methylchlorpyrifos $C_7H_7NO_3Cl_3PS$ [5598-13-0] HRBKVYFZANMGRE-UHFFFAOYSA-N | 4.1 2.5 2.9 3.3 $6.5\times10^{-1}$ 4.3 | | HSDB (2015) Mackay et al. (2006d) Suntio et al. (1988) Mackay and Shiu (1981) Hilal et al. (2008) Maniere et al. (2011) | V V V V Q ? | 12 241, 165 |
| dicapthon $C_8H_9NO_5ClPS$ [2463-84-5] OTKXWJHPGBRXCR-UHFFFAOYSA-N | $1.0\times10^2$ $4.2\times10^1$ $4.2\times10^1$ $4.4\times10^1$ 6.5 | | HSDB (2015) Mackay et al. (2006d) Suntio et al. (1988) Mackay and Shiu (1981) Hilal et al. (2008) | V V V V Q | 12 |
| chlorthion $C_8H_9ClNO_5PS$ [500-28-7] NZNRRXXETLSZRO-UHFFFAOYSA-N | $2.5\times10^2$ $2.4\times10^2$ | | HSDB (2015) MacBean (2012a) | V ? | |
| azamethiphos $C_9H_{10}ClN_2O_5PS$ [35575-96-3] VNKBTWQZTQIWDV-UHFFFAOYSA-N | $3.0\times10^5$ | | Ebert et al. (2023) | ? | 318 |
| isazophos $C_9H_{17}ClN_3O_3PS$ [42509-80-8] XRHGWAGWAHHFLF-UHFFFAOYSA-N | $1.9\times10^1$ $1.1\times10^2$ $7.2\times10^1$ | | HSDB (2015) Burkhard and Guth (1981) MacBean (2012a) | V V ? | |
| chlorpyrifos $C_9H_{11}Cl_3NO_3PS$ [2921-88-2] SBPBAQFWLVIOKP-UHFFFAOYSA-N | 1.8 2.1 $2.2\times10^{-1}$ 3.1 2.4 $9.2\times10^{-1}$ 1.7 $5.7\times10^{-1}$ $8.1\times10^{-1}$ $5.6\times10^{-3}$ 3.4 1.4 $1.0\times10^{-1}$ | 7800 | Muir et al. (2004) Muir et al. (2004) Cetin et al. (2006) Rice et al. (1997b) Fendinger and Glotfelty (1990) Mackay et al. (2006d) Siebers et al. (1994) Suntio et al. (1988) Glotfelty et al. (1987) Barcelo and Hennion (1997) HSDB (2015) Armbrust (2000) Goodarzi et al. (2010) | L L M M M V V V V X C C Q | 367 366 12 12 567 568, 569 |



Table A10.1: Phosphorus (C, H, O, N, Cl, Br, S, P) (...continued)

| Substance Formula (Trivial Name) [CAS Registry Number] InChIKey | $H_s^{cp}$ (at $T^\ominus$) $\left[\dfrac{\text{mol}}{\text{m}^3\,\text{Pa}}\right]$ | $\dfrac{\text{d}\ln H_s^{cp}}{\text{d}(1/T)}$ [K] | Reference | Type | Note |
|---|---|---|---|---|---|
| | $2.2\times10^{-1}$ | | Hilal et al. (2008) | Q | |
| | $2.5\times10^{2}$ | | Meylan and Howard (1991) | Q | |
| | 2.1 | | Maniere et al. (2011) | ? | 241, 165 |
| chlorphoxim $C_{12}H_{14}ClN_2O_3PS$ [14816-20-7] GQKRUMZWUHSLJF-NTCAYCPXSA-N | $>2.3\times10^{10}$ | | MacBean (2012a) | ? | |
| phosazetim $C_{14}H_{11}Cl_2N_2O_4PS$ [4104-14-7] XIBXUAZIZXDFTG-UHFFFAOYSA-N | $2.1\times10^{3}$ | | HSDB (2015) | Q | 99 |
| dialifor $C_{14}H_{17}ClNO_4PS_2$ [10311-84-9] MUMQYXACQUZOFP-UHFFFAOYSA-N | $5.5\times10^{1}$ 7.1 7.1 | | HSDB (2015) Mackay et al. (2006d) Suntio et al. (1988) | V V V | 12 |
| pyraclofos $C_{14}H_{18}ClN_2O_3PS$ [77458-01-6] QHGVXILFMXYDRS-UHFFFAOYSA-N | $2.3\times10^{4}$ | | Ebert et al. (2023) | ? | 318 |
| bromophos $C_8H_8BrCl_2O_3PS$ [2104-96-3] NYQDCVLCJXRDSK-UHFFFAOYSA-N | $1.0\times10^{-1}$ $1.1\times10^{-1}$ | | HSDB (2015) MacBean (2012a) | V ? | 12 |
| bromophos-ethyl $C_{10}H_{12}BrCl_2O_3PS$ [4824-78-6] KWGUFOITWDSNQY-UHFFFAOYSA-N | $6.2\times10^{-1}$ | | HSDB (2015) | Q | 99 |
| profenofos $C_{11}H_{15}BrClO_3PS$ [41198-08-7] QYMMJNLHFKGANY-UHFFFAOYSA-N | $4.5\times10^{2}$ $6.2\times10^{2}$ | | HSDB (2015) Mackay et al. (2006d) | V V | |
| iodofenphos $C_8H_8Cl_2IO_3PS$ [18181-70-9] LFVLUOAHQIVABZ-UHFFFAOYSA-N | 2.2 $>2.3\times10^{10}$ | | HSDB (2015) MacBean (2012a) | V ? | |



## A11  Organic species with other elements

### A11.1  Sodium (Na)

Table A11.1: Sodium (Na)

| Substance Formula (Trivial Name) [CAS Registry Number] InChIKey | $H_s^{cp}$ (at $T^{\ominus}$) $\left[\dfrac{\text{mol}}{\text{m}^3\,\text{Pa}}\right]$ | $\dfrac{\text{d}\ln H_s^{cp}}{\text{d}(1/T)}$ [K] | Reference | Type | Note |
|---|---|---|---|---|---|
| sesone $C_8H_7Cl_2NaO_5S$ (2,4-dichlorophenoxyethyl sulfate) [136-78-7] KISFEBPWFCGRGN-UHFFFAOYSA-M | $3.8\times10^5$ | | HSDB (2015) | Q | 99 |
| iodosulfuron-methyl-sodium $C_{14}H_{14}IN_5O_6SNa$ [144550-36-7] JUJFQMPKBJPSFZ-UHFFFAOYSA-M | $4.4\times10^{10}$ | | Maniere et al. (2011) | ? | 12, 165 |
| propoxycarbazone-sodium $C_{15}H_{17}N_4O_7NaS$ [181274-15-7] JRQGDDUXDKCWRF-UHFFFAOYSA-M | $>1.0\times10^{10}$ | | Maniere et al. (2011) | ? | 12, 165 |
| dioctyl sulfosuccinatesodium salt $C_{20}H_{37}NaO_7S$ (bis(2-ethylhexyl) sodium sulfosuccinate) [577-11-7] APSBXTVYXVQYAB-UHFFFAOYSA-M | $2.0\times10^6$ | | HSDB (2015) | Q | 99 |
| dodecylbenzenesulfonic acid sodium salt $C_{18}H_{29}NaO_3S$ (sodium dodecylbenzenesulfonate) [25155-30-0] BWUAQXVVJWAXHR-UHFFFAOYSA-M | $1.6\times10^2$ | | HSDB (2015) | Q | 99 |
| D&C yellow 10 $C_{20}H_{17}NO_8Na_2S_2$ [8004-92-0] NYMFWSWVZMTZQO-UHFFFAOYSA-L | $3.4\times10^{14}$ | | HSDB (2015) | Q | 99 |
| D&C yellow 8 $C_{20}H_{10}Na_2O_5$ (fluorescein sodium) [518-47-8] NJDNXYGOVLYJHP-UHFFFAOYSA-L | $3.5\times10^{10}$ | | HSDB (2015) | Q | 447 |
| D&C black 1 $C_{22}H_{14}N_6Na_2O_9S_2$ (amido black 10B) [1064-48-8] HKBVRFLHNUEVRO-DWTBGCDMSA-L | $8.2\times10^{25}$ | | HSDB (2015) | Q | 99 |



Table A11.1: Sodium (Na) (...continued)

| Substance Formula (Trivial Name) [CAS Registry Number] InChIKey | $H_s^{cp}$ (at $T^\ominus$) $\left[\dfrac{\mathrm{mol}}{\mathrm{m^3\,Pa}}\right]$ | $\dfrac{\mathrm{d}\ln H_s^{cp}}{\mathrm{d}(1/T)}$ [K] | Reference | Type | Note |
|---|---|---|---|---|---|
| D&C green 5 $C_{28}H_{20}N_2Na_2O_8S_2$ [4403-90-1] FPAYXBWMYIMERV-UHFFFAOYSA-L | $3.1\times10^{23}$ | | HSDB (2015) | Q | 447 |
| FD&C green 2 $C_{37}H_{34}N_2Na_2O_9S3$ [5141-20-8] DGOBMKYRQHEFGQ-UHFFFAOYSA-L | $7.0\times10^{30}$ | | HSDB (2015) | Q | 447 |



### A11.2 Aluminum (Al)

Table A11.2: Aluminum (Al)

| Substance Formula (Trivial Name) [CAS Registry Number] InChIKey | $H_s^{cp}$ (at $T^{\ominus}$) $\left[\dfrac{\text{mol}}{\text{m}^3\,\text{Pa}}\right]$ | $\dfrac{\text{d}\ln H_s^{cp}}{\text{d}(1/T)}$ [K] | Reference | Type | Note |
|---|---|---|---|---|---|
| fosetyl-aluminum $C_6H_{18}AlO_9P_3$ [39148-24-8] ZKZMJOFIHHZSRW-UHFFFAOYSA-K | $3.1\times10^9$ $>3.1\times10^9$ | | HSDB (2015) Maniere et al. (2011) | V ? | 12, 165 |





### A11.3 Silicon (Si)

Table A11.3: Silicon (Si)

| Substance Formula (Trivial Name) [CAS Registry Number] InChIKey | $H_s^{cp}$ (at $T^\ominus$) $\left[\dfrac{\text{mol}}{\text{m}^3\,\text{Pa}}\right]$ | $\dfrac{\text{d}\ln H_s^{cp}}{\text{d}(1/T)}$ [K] | Reference | Type | Note |
|---|---|---|---|---|---|
| tetramethylsilane $C_4H_{12}Si$ [75-76-3] CZDYPVPMEAXLPK-UHFFFAOYSA-N | $2.3\times10^{-6}$ $2.3\times10^{-6}$ $1.4\times10^{-3}$ $2.4\times10^{-6}$ | | Duchowicz et al. (2020) HSDB (2015) Duchowicz et al. (2020) Abraham et al. (1990) | V V Q ? | 186 |
| tetraethylsilane $C_8H_{20}Si$ [631-36-7] VCZQFJFZMMALHB-UHFFFAOYSA-N | $3.8\times10^{-6}$ | | Abraham et al. (1990) | ? | |
| trimethylsilanol $(CH_3)_3SiOH$ (TMS) [1066-40-6] AAPLIUHOKVUFCC-UHFFFAOYSA-N | $7.0\times10^{-2}$ $2.2\times10^{-1}$ | | Xu and Kropscott (2014) Mazzoni et al. (1997) | M V | |
| silicic acid $Si(OH)_4$ [10193-36-9] RMAQACBXLXPBSY-UHFFFAOYSA-N | $2.3\times10^{10}$ | 14000 | Plyasunov (2012) | M | 818 |
| dimethylsilanediol $C_2H_8O_2Si$ [1066-42-8] XCLIHDJZGPCUBT-UHFFFAOYSA-N | $2.8\times10^3$ $2.9\times10^{-1}$ | | Xu and Kropscott (2012) Mazzoni et al. (1997) | M V | 12 |
| tetramethyl silicate $C_4H_{12}O_4Si$ [681-84-5] LFQCEHFDDXELDD-UHFFFAOYSA-N | $1.5$ | | HSDB (2015) | Q | 99 |
| pentamethyldisiloxanol $C_5H_{16}O_2Si_2$ [56428-93-4] FGOLZCPMTWJPOU-UHFFFAOYSA-N | $7.3\times10^{-4}$ | | Mazzoni et al. (1997) | V | |
| tetraethyl silicate $C_8H_{20}O_4Si$ [78-10-4] BOTDANWDWHJENH-UHFFFAOYSA-N | $4.9\times10^{-1}$ | | HSDB (2015) | Q | 99 |
| trimethoxysilylpropyl methacrylate $C_{10}H_{20}O_5Si$ [2530-85-0] XDLMVUHYZWKMMD-UHFFFAOYSA-N | $3.3\times10^1$ | | HSDB (2015) | Q | 99 |



Table A11.3: Silicon (Si) (...continued)

| Substance Formula (Trivial Name) [CAS Registry Number] InChIKey | $H_s^{cp}$ (at $T^{\ominus}$) $\left[\dfrac{\mathrm{mol}}{\mathrm{m^3\,Pa}}\right]$ | $\dfrac{\mathrm{d}\ln H_s^{cp}}{\mathrm{d}(1/T)}$ [K] | Reference | Type | Note |
|---|---|---|---|---|---|
| hexamethyldisiloxane | $1.3\times10^{-6}$ | | Xu and Kropscott (2014) | M | |
| $C_6H_{18}OSi_2$ | $1.7\times10^{-4}$ | | Kochetkov et al. (2001) | M | 402, 330 |
| (L2) | $3.1\times10^{-4}$ | | Kochetkov et al. (2001) | M | 402, 331 |
| [107-46-0] | $7.7\times10^{-7}$ | | David et al. (2000) | M | 72 |
| UQEAIHBTYFGYIE-UHFFFAOYSA-N | $1.0\times10^{-6}$ | | Xu and Kropscott (2014) | V | |
| | $1.0\times10^{-6}$ | | Kochetkov et al. (2001) | V | |
| | $4.2\times10^{-6}$ | | Mazzoni et al. (1997) | V | |
| | $2.2\times10^{-4}$ | | Keshavarz et al. (2022) | Q | |
| | $1.9\times10^{-4}$ | | Duchowicz et al. (2020) | Q | |
| | $2.2\times10^{-4}$ | | Duchowicz et al. (2020) | ? | 185, 21 |
| octamethyltrisiloxane | $3.4\times10^{-7}$ | | Xu and Kropscott (2014) | M | |
| $C_8H_{24}O_2Si_3$ | $3.3\times10^{-6}$ | | Kochetkov et al. (2001) | M | 402, 330 |
| (L3) | $2.7\times10^{-6}$ | | Kochetkov et al. (2001) | M | 402, 331 |
| [107-51-7] | $2.8\times10^{-7}$ | | Xu and Kropscott (2014) | V | |
| CXQXSVUQTKDNFP-UHFFFAOYSA-N | $2.8\times10^{-7}$ | | Kochetkov et al. (2001) | V | |
| | $1.2\times10^{-6}$ | | Mazzoni et al. (1997) | V | |
| | $1.3\times10^{-5}$ | | Keshavarz et al. (2022) | Q | |
| | $2.5\times10^{-5}$ | | Duchowicz et al. (2020) | Q | |
| | $3.0\times10^{-6}$ | | Duchowicz et al. (2020) | ? | 185, 21 |
| decamethyltetrasiloxane | $1.4\times10^{-7}$ | | Xu and Kropscott (2014) | M | |
| $C_{10}H_{30}O_3Si_4$ | $5.8\times10^{-7}$ | | Kochetkov et al. (2001) | M | 402, 330 |
| (L4) | $3.7\times10^{-7}$ | | Xu and Kropscott (2014) | V | |
| [141-62-8] | $4.3\times10^{-7}$ | | Kochetkov et al. (2001) | V | |
| YFCGDEUVHLPRCZ-UHFFFAOYSA-N | $3.1\times10^{-7}$ | | Mazzoni et al. (1997) | V | |
| | $8.1\times10^{-7}$ | | Keshavarz et al. (2022) | Q | |
| | $3.3\times10^{-6}$ | | Duchowicz et al. (2020) | Q | 184 |
| | $5.8\times10^{-7}$ | | Duchowicz et al. (2020) | ? | 185, 21 |
| dodecamethylpentasiloxane | $8.7\times10^{-8}$ | | Mazzoni et al. (1997) | V | |
| $C_{12}H_{36}O_4Si_5$ | | | | | |
| (L5) | | | | | |
| [141-63-9] | | | | | |
| FBZANXDWQAVSTQ-UHFFFAOYSA-N | | | | | |
| silthiofam | 1.9 | | Maniere et al. (2011) | ? | 241, 165 |
| $C_{13}H_{21}NOSSi$ | | | | | |
| [175217-20-6] | | | | | |
| MXMXHPPIGKYTAR-UHFFFAOYSA-N | | | | | |
| tetradecamethylhexasiloxane | $2.7\times10^{-8}$ | | Mazzoni et al. (1997) | V | |
| $C_{14}H_{42}O_5Si_6$ | | | | | |
| (L6) | | | | | |
| [107-52-8] | | | | | |
| ADANNTOYRVPQLJ-UHFFFAOYSA-N | | | | | |



Table A11.3: Silicon (Si) (...continued)

| Substance<br>Formula<br>(Trivial Name)<br>[CAS Registry Number]<br>InChIKey | $H_s^{cp}$<br>(at $T^{\ominus}$)<br>$\left[\dfrac{\text{mol}}{\text{m}^3\,\text{Pa}}\right]$ | $\dfrac{\text{d}\ln H_s^{cp}}{\text{d}(1/T)}$<br><br>[K] | Reference | Type | Note |
|---|---|---|---|---|---|
| hexadecamethylheptasiloxane<br>$C_{16}H_{48}O_6Si_7$<br>(L7)<br>[541-01-5]<br>NFVSFLUJRHRSJG-UHFFFAOYSA-N | $7.6\times10^{-9}$ | | Mazzoni et al. (1997) | V | |
| octadecamethyloctasiloxane<br>$C_{18}H_{54}O_7Si_8$<br>(L8)<br>[556-69-4]<br>VWGDPBQTSZDFMX-UHFFFAOYSA-N | $3.3\times10^{-9}$ | | Mazzoni et al. (1997) | V | |
| hexamethylcyclotrisiloxane<br>$C_6H_{18}O_3Si_3$<br>(D3)<br>[541-05-9]<br>HTDJPCNNEPUOOQ-UHFFFAOYSA-N | $5.6\times10^{-6}$<br>$6.4\times10^{-6}$<br>$1.6\times10^{-5}$<br>$2.5\times10^{-6}$ | | Mazzoni et al. (1997)<br>Alton and Browne (2022)<br>Alton and Browne (2022)<br>Alton and Browne (2022) | V<br>Q<br>Q<br>Q | <br>819<br>820<br>821 |
| octamethylcyclotetrasiloxane<br>$C_8H_{24}O_4Si_4$<br>(D4)<br>[556-67-2]<br>HMMGMWAXVFQUOA-UHFFFAOYSA-N | $7.3\times10^{-7}$<br>$8.3\times10^{-7}$<br>$1.7\times10^{-5}$<br>$1.7\times10^{-5}$<br>$1.2\times10^{-4}$<br>$1.5\times10^{-6}$<br>$1.6\times10^{-6}$<br>$8.3\times10^{-7}$<br>$2.7\times10^{-6}$<br>$5.4\times10^{-7}$<br>$1.3\times10^{-6}$<br>$1.1\times10^{-6}$<br>$1.3\times10^{-5}$<br>$4.2\times10^{-5}$<br>$8.4\times10^{-5}$ | | Xu and Kropscott (2014)<br>Xu and Kropscott (2012)<br>Kochetkov et al. (2001)<br>Kochetkov et al. (2001)<br>Hamelink et al. (1996)<br>Xu and Kropscott (2014)<br>Kochetkov et al. (2001)<br>Mazzoni et al. (1997)<br>Hamelink et al. (1996)<br>Alton and Browne (2022)<br>Alton and Browne (2022)<br>Alton and Browne (2022)<br>Keshavarz et al. (2022)<br>Duchowicz et al. (2020)<br>Duchowicz et al. (2020) | M<br>M<br>M<br>M<br>M<br>V<br>V<br>V<br>V<br>Q<br>Q<br>Q<br>Q<br>Q<br>? | <br>87<br>297, 330<br>297, 331<br>12<br><br><br><br>12<br>819<br>820<br>821<br><br><br>185, 21 |
| decamethylcyclopentasiloxane<br>$C_{10}H_{30}O_5Si_5$<br>(D5)<br>[541-02-6]<br>XMSXQFUHVRWGNA-UHFFFAOYSA-N | $2.8\times10^{-7}$<br>$3.0\times10^{-7}$<br>$3.4\times10^{-5}$<br>$3.1\times10^{-5}$<br>$7.4\times10^{-5}$<br>$2.3\times10^{-6}$<br>$2.2\times10^{-6}$<br>$1.5\times10^{-6}$<br>$2.3\times10^{-7}$<br>$1.2\times10^{-6}$<br>$3.1\times10^{-6}$ | | Xu and Kropscott (2014)<br>Xu and Kropscott (2012)<br>Kochetkov et al. (2001)<br>Kochetkov et al. (2001)<br>David et al. (2000)<br>Xu and Kropscott (2014)<br>Kochetkov et al. (2001)<br>Mazzoni et al. (1997)<br>Alton and Browne (2022)<br>Alton and Browne (2022)<br>Alton and Browne (2022) | M<br>M<br>M<br>M<br>M<br>V<br>V<br>V<br>Q<br>Q<br>Q | <br><br>372, 330<br>372, 331<br>72<br><br><br><br>819<br>820<br>821 |



Table A11.3: Silicon (Si) (...continued)

| Substance Formula (Trivial Name) [CAS Registry Number] InChIKey | $H_s^{cp}$ (at $T^{\ominus}$) $\left[\dfrac{\text{mol}}{\text{m}^3\,\text{Pa}}\right]$ | $\dfrac{\text{d}\ln H_s^{cp}}{\text{d}(1/T)}$ [K] | Reference | Type | Note |
|---|---|---|---|---|---|
| dodecamethylcyclohexasiloxane $C_{12}H_{36}O_6Si_6$ (D6) [540-97-6] IUMSDRXLFWAGNT-UHFFFAOYSA-N | $4.0\times10^{-7}$ $6.8\times10^{-5}$ $1.5\times10^{-4}$ $3.9\times10^{-6}$ | | Xu and Kropscott (2012) Kochetkov et al. (2001) Kochetkov et al. (2001) Kochetkov et al. (2001) | M M M V | 79 372, 330 372, 331 |
| tetramethyldisiloxane-1,3-diol $C_4H_{14}O_3Si_2$ [1118-15-6] PFEAZKFNWPIFCV-UHFFFAOYSA-N | $1.8\times10^{-1}$ | | Mazzoni et al. (1997) | V | |
| hexamethyltrisiloxane-1,5-diol $C_6H_{20}O_4Si_3$ [3663-50-1] XYBQTTAROZGWOZ-UHFFFAOYSA-N | $3.4\times10^{-3}$ | | Mazzoni et al. (1997) | V | |
| octamethyltetrasiloxane-1,7-diol $C_8H_{26}O_5Si_4$ [3081-07-0] VERNMKKMBJGSQB-UHFFFAOYSA-N | $2.7\times10^{-3}$ | | Mazzoni et al. (1997) | V | |
| pentamethylcyclotrisiloxanol $C_5H_{16}O_4Si_3$ (D3OH) [106916-50-1] OGNYSZJXUXVWRE-UHFFFAOYSA-N | $1.1\times10^{-3}$ $1.2\times10^{-1}$ $2.7\times10^{-2}$ $1.5\times10^{-3}$ | | Mazzoni et al. (1997) Alton and Browne (2022) Alton and Browne (2022) Alton and Browne (2022) | V Q Q Q | 819 820 821 |
| heptamethylcyclotetrasiloxanol $C_7H_{22}O_5Si_4$ (D4OH) [5290-02-8] MLGUQSFCJNMKPT-UHFFFAOYSA-N | $2.3\times10^{-4}$ $1.1\times10^{-2}$ $1.1\times10^{-2}$ $1.3\times10^{-5}$ | | Mazzoni et al. (1997) Alton and Browne (2022) Alton and Browne (2022) Alton and Browne (2022) | V Q Q Q | 819 820 821 |
| nonamethylcyclopentasiloxanol $C_9H_{28}O_6Si_5$ (D5OH) [5290-04-0] DKTKKERLEMHMHS-UHFFFAOYSA-N | $7.0\times10^{-5}$ $4.6\times10^{-3}$ $2.0\times10^{-2}$ $4.4\times10^{-5}$ | | Mazzoni et al. (1997) Alton and Browne (2022) Alton and Browne (2022) Alton and Browne (2022) | V Q Q Q | 819 820 821 |
| D3FormEst $C_6H_{16}O_5Si_3$ TZIXLALHTGGYNZ-UHFFFAOYSA-N | $2.8\times10^{-4}$ $3.2\times10^{-4}$ $1.7\times10^{-3}$ | | Alton and Browne (2022) Alton and Browne (2022) Alton and Browne (2022) | Q Q Q | 819 820 821 |
| D3OHFormEst $C_5H_{14}O_6Si_3$ RMJPVGNUQFKGPC-UHFFFAOYSA-N | $5.4$ $9.9$ $1.2\times10^{-2}$ | | Alton and Browne (2022) Alton and Browne (2022) Alton and Browne (2022) | Q Q Q | 819 820 821 |
| D3Hydroperoxide $C_6H_{18}O_5Si_3$ JKDUHABXPCJOJB-UHFFFAOYSA-N | $6.9\times10^{-2}$ $1.9\times10^{-3}$ $8.9\times10^{-6}$ | | Alton and Browne (2022) Alton and Browne (2022) Alton and Browne (2022) | Q Q Q | 819 820 821 |





Table A11.3: Silicon (Si) (...continued)

| Substance Formula (Trivial Name) [CAS Registry Number] InChIKey | $H_s^{cp}$ (at $T^\ominus$) $\left[\dfrac{\text{mol}}{\text{m}^3\,\text{Pa}}\right]$ | $\dfrac{\text{d}\ln H_s^{cp}}{\text{d}(1/T)}$ [K] | Reference | Type | Note |
|---|---|---|---|---|---|
| D3EtherHydroperoxide $C_6H_{18}O_6Si_3$ LHGAYMWHCKTFJV-UHFFFAOYSA-N | $8.4\times10^{-1}$ $1.9\times10^{-2}$ $1.7\times10^{-2}$ | | Alton and Browne (2022) Alton and Browne (2022) Alton and Browne (2022) | Q Q Q | 819 820 821 |
| D3FormEst2 $C_6H_{14}O_7Si_3$ WKTLUKWDJJZJCN-UHFFFAOYSA-N | $1.2\times10^{-2}$ $9.2\times10^{-3}$ $2.0\times10^{-2}$ | | Alton and Browne (2022) Alton and Browne (2022) Alton and Browne (2022) | Q Q Q | 819 820 821 |
| D3OH2O1 $C_4H_{14}O_5Si_3$ UGEQNRIYBONGCP-UHFFFAOYSA-N | $2.4\times10^3$ $2.9$ $1.8\times10^{-1}$ | | Alton and Browne (2022) Alton and Browne (2022) Alton and Browne (2022) | Q Q Q | 819 820 821 |
| D4FormEst $C_8H_{22}O_6Si_4$ PWVQOGDRWKITJQ-UHFFFAOYSA-N | $2.4\times10^{-5}$ $8.5\times10^{-5}$ $8.8\times10^{-5}$ | | Alton and Browne (2022) Alton and Browne (2022) Alton and Browne (2022) | Q Q Q | 819 820 821 |
| D4OHFormEst $C_7H_{20}O_7Si_4$ AXBVNNKTZSENOJ-UHFFFAOYSA-N | $4.6\times10^{-1}$ $3.1$ $1.2\times10^{-4}$ | | Alton and Browne (2022) Alton and Browne (2022) Alton and Browne (2022) | Q Q Q | 819 820 821 |
| 1,1,1,3,5,5,5- heptamethyltrisiloxane $C_7H_{22}O_2Si_3$ [1873-88-7] QNWOFLWXQGHSRH-UHFFFAOYSA-N | $1.1\times10^{-7}$ | | Ebert et al. (2023) | ? | 318 |
| D4Hydroperoxide $C_8H_{24}O_6Si_4$ IEOAKTNTPADFJB-UHFFFAOYSA-N | $5.8\times10^{-3}$ $4.4\times10^{-4}$ $1.8\times10^{-6}$ | | Alton and Browne (2022) Alton and Browne (2022) Alton and Browne (2022) | Q Q Q | 819 820 821 |
| D4EtherHydroperoxide $C_8H_{24}O_7Si_4$ BKBHFCQPJIYNBR-UHFFFAOYSA-N | $7.1\times10^{-2}$ $4.5\times10^{-3}$ $1.5\times10^{-5}$ | | Alton and Browne (2022) Alton and Browne (2022) Alton and Browne (2022) | Q Q Q | 819 820 821 |
| D4FormEst2 $C_8H_{20}O_8Si_4$ WJOJQLFAIHHNFX-UHFFFAOYSA-N | $1.0\times10^{-3}$ $1.8\times10^{-3}$ $2.9\times10^{-4}$ | | Alton and Browne (2022) Alton and Browne (2022) Alton and Browne (2022) | Q Q Q | 819 820 821 |
| D4OH2 $C_6H_{20}O_6Si_4$ ONAQLRRZBYFKLA-UHFFFAOYSA-N | $2.1\times10^2$ $1.0$ $8.3\times10^{-5}$ | | Alton and Browne (2022) Alton and Browne (2022) Alton and Browne (2022) | Q Q Q | 819 820 821 |
| D5FormEst $C_{10}H_{28}O_7Si_5$ IGERVNZEGVJPED-UHFFFAOYSA-N | $1.0\times10^{-5}$ $1.2\times10^{-4}$ $6.4\times10^{-5}$ | | Alton and Browne (2022) Alton and Browne (2022) Alton and Browne (2022) | Q Q Q | 819 820 821 |
| glycidoxypropyltrimethoxysilane $C_9H_{20}O_5Si$ [2530-83-8] BPSIOYPQMFLKFR-UHFFFAOYSA-N | $1.4\times10^2$ | | Ebert et al. (2023) | ? | 318 |



Table A11.3: Silicon (Si) (...continued)

| Substance Formula (Trivial Name) [CAS Registry Number] InChIKey | $H_s^{cp}$ (at $T^{\ominus}$) $\left[\dfrac{\mathrm{mol}}{\mathrm{m}^3\,\mathrm{Pa}}\right]$ | $\dfrac{\mathrm{d}\ln H_s^{cp}}{\mathrm{d}(1/T)}$ [K] | Reference | Type | Note |
|---|---|---|---|---|---|
| D5OHFormEst $C_9H_{26}O_8Si_5$ KNIXZSAQQHGMJX-UHFFFAOYSA-N | $2.0\times10^{-1}$ 5.6 $4.7\times10^{-5}$ | | Alton and Browne (2022) Alton and Browne (2022) Alton and Browne (2022) | Q Q Q | 819 820 821 |
| D5Hydroperoxide $C_{10}H_{30}O_7Si_5$ VGMLSGIAATVHBV-UHFFFAOYSA-N | $2.5\times10^{-3}$ $6.3\times10^{-4}$ $1.1\times10^{-6}$ | | Alton and Browne (2022) Alton and Browne (2022) Alton and Browne (2022) | Q Q Q | 819 820 821 |
| D5EtherHydroperoxide $C_{10}H_{30}O_8Si_5$ GSAPDUDDTPAQSH-UHFFFAOYSA-N | $3.1\times10^{-2}$ $6.8\times10^{-3}$ $1.1\times10^{-5}$ | | Alton and Browne (2022) Alton and Browne (2022) Alton and Browne (2022) | Q Q Q | 819 820 821 |
| D5FormEst2 $C_{10}H_{26}O_9Si_5$ QFORSIVEJZTNKQ-UHFFFAOYSA-N | $4.5\times10^{-4}$ $2.5\times10^{-3}$ $2.0\times10^{-4}$ | | Alton and Browne (2022) Alton and Browne (2022) Alton and Browne (2022) | Q Q Q | 819 820 821 |
| isobutyltriethoxysilane $C_{10}H_{24}O_3Si$ [17980-47-1] ALVYUZIFSCKIFP-UHFFFAOYSA-N | $5.5\times10^{-3}$ | | Ebert et al. (2023) | ? | 316 |
| methyltris(trimethylsiloxy)silane $C_{10}H_{30}O_3Si_4$ [17928-28-8] RGMZNZABJYWAEC-UHFFFAOYSA-N | $2.9\times10^{-8}$ | | Ebert et al. (2023) | ? | 316 |
| vinyltris(2-methoxyethoxy)silane $C_{11}H_{24}O_6Si$ [1067-53-4] WOXXJEVNDJOOLV-UHFFFAOYSA-N | $5.9\times10^{2}$ | | Ebert et al. (2023) | ? | 318 |
| tetrakis(trimethylsiloxy)silane $C_{12}H_{36}O_4Si_5$ [3555-47-3] VNRWTCZXQWOWIG-UHFFFAOYSA-N | $2.6\times10^{-8}$ | | Ebert et al. (2023) | ? | 318 |
| 2-(3,4-epoxycyclohexyl) ethyltriethoxysilane $C_{14}H_{28}O_4Si$ [10217-34-2] UDUKMRHNZZLJRB-UHFFFAOYSA-N | 9.9 | | Ebert et al. (2023) | ? | 318 |
| D5OH2 $C_8H_{26}O_7Si_5$ AGIASGIZCMBNMT-UHFFFAOYSA-N | $8.9\times10^{1}$ 2.5 $1.3\times10^{-4}$ | | Alton and Browne (2022) Alton and Browne (2022) Alton and Browne (2022) | Q Q Q | 819 820 821 |
| hexamethyldisilazane $C_6H_{19}NSi_2$ [999-97-3] FFUAGWLWBBFQJT-UHFFFAOYSA-N | $1.1\times10^{-1}$ | | HSDB (2015) | Q | 99 |



Table A11.3: Silicon (Si) (... continued)

| Substance Formula (Trivial Name) [CAS Registry Number] InChIKey | $H_s^{cp}$ (at $T^{\ominus}$) $\left[\dfrac{\mathrm{mol}}{\mathrm{m}^3\,\mathrm{Pa}}\right]$ | $\dfrac{\mathrm{d}\ln H_s^{cp}}{\mathrm{d}(1/T)}$ [K] | Reference | Type | Note |
|---|---|---|---|---|---|
| D3Organonitrate $C_6H_{17}NO_6Si_3$ WGZJMUDCTAXYSF-UHFFFAOYSA-N | $2.4\times10^{-3}$ $1.6\times10^{-4}$ $1.0\times10^{-4}$ | | Alton and Browne (2022) Alton and Browne (2022) Alton and Browne (2022) | Q Q Q | 819 820 821 |
| D3EtherOrganonitrate $C_6H_{17}NO_7Si_3$ CJWFNCZCCFYEKP-UHFFFAOYSA-N | $2.9\times10^{-2}$ $1.7\times10^{-3}$ $1.4\times10^{-3}$ | | Alton and Browne (2022) Alton and Browne (2022) Alton and Browne (2022) | Q Q Q | 819 820 821 |
| D4Organonitrate $C_8H_{23}NO_7Si_4$ CYMCLGYPZZKOOI-UHFFFAOYSA-N | $2.0\times10^{-4}$ $4.1\times10^{-5}$ $7.5\times10^{-6}$ | | Alton and Browne (2022) Alton and Browne (2022) Alton and Browne (2022) | Q Q Q | 819 820 821 |
| D4EtherOrganonitrate $C_8H_{23}NO_8Si_4$ XHFUWVKWVPBCDO-UHFFFAOYSA-N | $2.4\times10^{-3}$ $3.8\times10^{-4}$ $1.6\times10^{-5}$ | | Alton and Browne (2022) Alton and Browne (2022) Alton and Browne (2022) | Q Q Q | 819 820 821 |
| (3-aminopropyl)triethoxysilane $C_9H_{23}NO_3Si$ [919-30-2] WYTZZXDRDKSJID-UHFFFAOYSA-N | $1.2\times10^{2}$ | | Ebert et al. (2023) | ? | 318 |
| D5Organonitrate $C_{10}H_{29}NO_8Si_5$ KWHVDACCYNMOAR-UHFFFAOYSA-N | $8.7\times10^{-5}$ $4.9\times10^{-5}$ $1.2\times10^{-5}$ | | Alton and Browne (2022) Alton and Browne (2022) Alton and Browne (2022) | Q Q Q | 819 820 821 |
| D5EtherOrganonitrate $C_{10}H_{29}NO_9Si_5$ BQINUVFPZWRRBW-UHFFFAOYSA-N | $1.1\times10^{-3}$ $5.7\times10^{-4}$ $2.9\times10^{-5}$ | | Alton and Browne (2022) Alton and Browne (2022) Alton and Browne (2022) | Q Q Q | 819 820 821 |
| silafluofen $C_{25}H_{29}FO_2Si$ [105024-66-6] HPYNBECUCCGGPA-UHFFFAOYSA-N | $4.0\times10^{-1}$ | | Ebert et al. (2023) | ? | 318 |
| flusilazole $C_{16}H_{15}F_2N_3Si$ [85509-19-9] FQKUGOMFVDPBIZ-UHFFFAOYSA-N | $2.7\times10^{1}$ $4.4\times10^{2}$ | | Barcelo and Hennion (1997) Goodarzi et al. (2010) | X Q | 567 568, 571 |
| simeconazole $C_{14}H_{20}FN_3OSi$ [149508-90-7] YABFPHSQTSFWQB-UHFFFAOYSA-N | $4.0\times10^{3}$ | | Ebert et al. (2023) | ? | 318 |
| dichloromethylsilane $CH_4Cl_2Si$ (methyldichlorosilane) [75-54-7] NWKBSEBOBPHMKL-UHFFFAOYSA-N | $7.6\times10^{-4}$ | | HSDB (2015) | Q | 99 |



Table A11.3: Silicon (Si) (... continued)

| Substance Formula (Trivial Name) [CAS Registry Number] InChIKey | $H_s^{cp}$ (at $T^\ominus$) $\left[ \dfrac{\mathrm{mol}}{\mathrm{m}^3\,\mathrm{Pa}} \right]$ | $\dfrac{\mathrm{d}\ln H_s^{cp}}{\mathrm{d}(1/T)}$ [K] | Reference | Type | Note |
|---|---|---|---|---|---|
| etacelasil C$_{11}$H$_{25}$O$_6$ClSi [37894-46-5] SLZWEMYSYKOWCG-UHFFFAOYSA-N | $2.9 \times 10^3$ | | MacBean (2012a) | ? | |





### A11.4 Calcium (Ca)

Table A11.4: Calcium (Ca)

| Substance Formula (Trivial Name) [CAS Registry Number] InChIKey | $H_s^{cp}$ (at $T^{\ominus}$) $\left[\dfrac{\text{mol}}{\text{m}^3\,\text{Pa}}\right]$ | $\dfrac{\text{d}\ln H_s^{cp}}{\text{d}(1/T)}$ [K] | Reference | Type | Note |
|---|---|---|---|---|---|
| prohexadione-calcium $C_{20}H_{22}O_{10}Ca$ [127277-53-6] VQIYODJWCVPJRC-UHFFFAOYSA-N | $5.2 \times 10^4$ | | Maniere et al. (2011) | ? | 241, 165 |



## A11.5   Zinc (Zn)

Table A11.5: Zinc (Zn)

| Substance Formula (Trivial Name) [CAS Registry Number] InChIKey | $H_s^{cp}$ (at $T^\ominus$) $\left[\dfrac{\mathrm{mol}}{\mathrm{m^3\,Pa}}\right]$ | $\dfrac{\mathrm{d}\ln H_s^{cp}}{\mathrm{d}(1/T)}$ [K] | Reference | Type | Note |
|---|---|---|---|---|---|
| zineb $C_4H_6N_2S_4Zn$ [12122-67-7] AMHNZOICSMBGDH-UHFFFAOYSA-L | $2.7\times10^3$ $>3.7\times10^3$ $>1.9\times10^2$ | | Mackay et al. (2006d) MacBean (2012b) Maniere et al. (2011) | V X ? | 350 12, 165 |
| ziram $C_6H_{12}N_2S_4Zn$ [137-30-4] DUBNHZYBDBBJHD-UHFFFAOYSA-L | $1.6\times10^4$ $2.1\times10^5$ $1.8\times10^2$ | | HSDB (2015) Mackay et al. (2006d) Maniere et al. (2011) | V V ? | 165 |
| mancozeb $C_8H_{12}MnN_4S_8Zn$ [8018-01-7] CHNQZRKUZPNOOH-UHFFFAOYSA-J | $1.6\times10^1$ | | Maniere et al. (2011) | ? | 12, 165 |



### A11.6 Arsenic (Sn)

Table A11.6: Arsenic (Sn)

| Substance Formula (Trivial Name) [CAS Registry Number] InChIKey | $H_s^{cp}$ (at $T^{\ominus}$) $\left[\dfrac{\mathrm{mol}}{\mathrm{m^3\,Pa}}\right]$ | $\dfrac{\mathrm{d}\ln H_s^{cp}}{\mathrm{d}(1/T)}$ [K] | Reference | Type | Note |
|---|---|---|---|---|---|
| methylarsine $CH_5As$ [593-52-2] IDDBICIFODFKQO-UHFFFAOYSA-N | $9.0{\times}10^{-6}$ | | Ebert et al. (2023) | ? | 318 |
| ethylarsine $C_2H_7As$ [593-59-9] OSAGMAAQMRQLLW-UHFFFAOYSA-N | $1.5{\times}10^{-5}$ | | Ebert et al. (2023) | ? | 316 |
| diethyl arsine $C_4H_{11}As$ [692-42-2] JZCIYTSNUPIOMK-UHFFFAOYSA-N | $2.2{\times}10^{-5}$ | | HSDB (2015) | Q | 99 |
| phenylarsine oxide $C_6H_5AsO$ [637-03-6] BQVCCPGCDUSGOE-UHFFFAOYSA-N | $9.0{\times}10^{-1}$ | | Bartelt-Hunt et al. (2008) | ? | 21 |
| diphenylarsanylformonitrile $C_{13}H_{10}AsN$ [23525-22-6] BDHNJKLLVSRGDK-UHFFFAOYSA-N | $1.1{\times}10^{3}$ | | Ebert et al. (2023) | ? | 318 |
| dichloro(methyl)arsane $CH_3AsCl_2$ [593-89-5] VXRMBBLRHSRVDK-UHFFFAOYSA-N | $5.1{\times}10^{-3}$ | | Ebert et al. (2023) | ? | 318 |
| lewisite $C_2H_2AsCl_3$ [541-25-3] GIKLTQKNOXNBNY-OWOJBTEDSA-N | $4.5{\times}10^{-2}$ $3.1{\times}10^{-2}$ $5.2{\times}10^{-3}$ $3.7{\times}10^{-2}$ | | Duchowicz et al. (2020) HSDB (2015) Duchowicz et al. (2020) Bartelt-Hunt et al. (2008) | V V Q ? | 186 21 |
| diphenylchloroarsine $C_{12}H_{10}AsCl$ [712-48-1] YHHKGKCOLGRKKB-UHFFFAOYSA-N | $6.1{\times}10^{1}$ | | Ebert et al. (2023) | ? | 318 |
| lewisite oxide $C_2H_2AsClO$ [3088-37-7] MVCVAGFCWDFQQX-OWOJBTEDSA-N | $5.2{\times}10^{-3}$ | | Bartelt-Hunt et al. (2008) | ? | 21 |



Table A11.6: Arsenic (Sn) (. . . continued)

| Substance Formula (Trivial Name) [CAS Registry Number] InChIKey | $H_s^{cp}$ (at $T^{\ominus}$) $\left[\dfrac{\mathrm{mol}}{\mathrm{m}^3\,\mathrm{Pa}}\right]$ | $\dfrac{\mathrm{d}\ln H_s^{cp}}{\mathrm{d}(1/T)}$ [K] | Reference | Type | Note |
|---|---|---|---|---|---|
| phenyldichloroarsine $C_6H_5AsCl_2$ [696-28-6] UDHDFEGCOJAVRE-UHFFFAOYSA-N | $3.3\times10^{-1}$ | | HSDB (2015) | Q | 99 |
| adamsite $C_{12}H_9AsClN$ [578-94-9] PBNSPNYJYOYWTA-UHFFFAOYSA-N | $3.0\times10^2$ | | HSDB (2015) | Q | 99 |



### A11.7 Selenium (Se)

Table A11.7: Selenium (Se)

| Substance Formula (Trivial Name) [CAS Registry Number] InChIKey | $H_s^{cp}$ (at $T^{\ominus}$) $\left[\dfrac{\text{mol}}{\text{m}^3\,\text{Pa}}\right]$ | $\dfrac{\text{d}\ln H_s^{cp}}{\text{d}(1/T)}$ [K] | Reference | Type | Note |
|---|---|---|---|---|---|
| dimethyl selenide C$_2$H$_6$Se [593-79-3] RVIXKDRPFPUUOO-UHFFFAOYSA-N | $1.2\times10^{-2}$ $4.3\times10^{-3}$ $5.7\times10^{-3}$ | | Keshavarz et al. (2022) Duchowicz et al. (2020) Duchowicz et al. (2020) | Q Q ? | 185, 21 |
| dimethyl diselenide C$_2$H$_6$Se$_2$ [7101-31-7] VLXBWPOEOIIREY-UHFFFAOYSA-N | $1.4\times10^{-3}$ | | Ebert et al. (2023) | ? | 318 |
| 2-amino-4-(methylselenyl)butyric acid C$_5$H$_{11}$NO$_2$Se (selenium methionine) [1464-42-2] RJFAYQIBOAGBLC-UHFFFAOYSA-N | $2.9\times10^{5}$ | | HSDB (2015) | Q | 99 |



**A11.8   Tin (Sn)**

Table A11.8: Tin (Sn)

| Substance<br>Formula<br>(Trivial Name)<br>[CAS Registry Number]<br>InChIKey | $H_s^{cp}$<br>(at $T^{\ominus}$)<br><br>$\left[\dfrac{\text{mol}}{\text{m}^3\,\text{Pa}}\right]$ | $\dfrac{\text{d}\ln H_s^{cp}}{\text{d}(1/T)}$<br><br>[K] | Reference | Type | Note |
|---|---|---|---|---|---|
| tetramethylstannane<br>$C_4H_{12}Sn$<br>(tetramethyltin)<br>[594-27-4]<br>VXKWYPOMXBVZSJ-UHFFFAOYSA-N | $9.4\times10^{-6}$<br>$9.7\times10^{-6}$<br>$1.2\times10^{-5}$ | 3800 | Abraham and Nasehzadeh (1981)<br>Abraham et al. (1990)<br>Abraham (1979) | M<br>?<br>? | |
| tetraethylstannane<br>$C_8H_{20}Sn$<br>(tetraethyltin)<br>[597-64-8]<br>RWWNQEOPUOCKGR-UHFFFAOYSA-N | $1.6\times10^{-5}$<br>$6.1\times10^{-6}$<br>$5.7\times10^{-6}$<br>$1.1\times10^{-5}$ | <br><br>6100 | HSDB (2015)<br>Abraham et al. (1990)<br>Abraham and Nasehzadeh (1981)<br>Abraham (1979) | Q<br>?<br>?<br>? | 99<br><br>822 |
| tetrabutylstannane<br>$C_{16}H_{36}Sn$<br>(tetra-butyl tin)<br>[1461-25-2]<br>AFCAKJKUYFLYFK-UHFFFAOYSA-N | $1.6\times10^{-6}$ | | HSDB (2015) | Q | 99 |
| triphenyltin hydroxide<br>$C_{18}H_{16}OSn$<br>[76-87-9]<br>BFWMWWXRWVJXSE-UHFFFAOYSA-M | $2.3\times10^{1}$<br>$3.8\times10^{-1}$ | | Duchowicz et al. (2020)<br>Duchowicz et al. (2020) | V<br>Q | 186 |
| hexabutyldistannoxane<br>$C_{24}H_{54}OSn_2$<br>(bis(tributyltin)oxide)<br>[56-35-9]<br>APQHKWPGGHMYKJ-UHFFFAOYSA-N | $7.6\times10^{1}$ | | HSDB (2015) | V | |
| hexakis(2-methyl-2-<br>phenylpropyl)distannoxane<br>$C_{60}H_{78}OSn_2$<br>(fenbutatin oxide)<br>[13356-08-6]<br>HOXINJBQVZWYGZ-UHFFFAOYSA-N | $4.9\times10^{3}$ | | HSDB (2015) | V | |
| fentin acetate<br>$C_{20}H_{18}O_2Sn$<br>[900-95-8]<br>WDQNIWFZKXZFAY-UHFFFAOYSA-M | $2.0\times10^{3}$ | | Ebert et al. (2023) | ? | 365 |
| 1-(tricyclohexylstannyl)1H-1,2,4-<br>triazole<br>$C_{20}H_{35}N_3Sn$<br>(azocyclotin)<br>[41083-11-8]<br>ONHBDDJJTDTLIR-UHFFFAOYSA-N | $4.6\times10^{6}$<br><br>$4.6\times10^{6}$<br>1.2 | | Duchowicz et al. (2020)<br><br>HSDB (2015)<br>Duchowicz et al. (2020) | V<br><br>V<br>Q | 186 |



### A11.9 Mercury (Hg)

Table A11.9: Mercury (Hg)

| Substance<br>Formula<br>(Trivial Name)<br>[CAS Registry Number]<br>InChIKey | $H_s^{cp}$<br>(at $T^{\ominus}$)<br>$\left[\dfrac{\mathrm{mol}}{\mathrm{m^3\,Pa}}\right]$ | $\dfrac{\mathrm{d}\ln H_s^{cp}}{\mathrm{d}(1/T)}$<br><br>[K] | Reference | Type | Note |
|---|---|---|---|---|---|
| dimethylmercury<br>$C_2H_6Hg$<br>[593-74-8]<br>ATZBPOVXVPIOMR-UHFFFAOYSA-N | $1.3\times10^{-3}$<br>$2.1\times10^{-3}$<br>$1.3\times10^{-3}$<br>$1.0\times10^{-3}$<br>$1.5\times10^{-3}$<br>$1.3\times10^{-3}$<br>$3.1\times10^{-3}$ | 2700<br><br>2700<br>3000<br><br>2700<br> | Talmi and Mesmer (1975)<br>Abraham et al. (2008)<br>WHO (1990)<br>Abraham et al. (2008)<br>Schroeder and Munthe (1998)<br>Schroeder and Munthe (1998)<br>Iverfeldt and Persson (1985) | M<br>C<br>C<br>Q<br>?<br>?<br>? | <br><br><br>218<br>21<br>21<br>219 |
| diethylmercury<br>$C_4H_{10}Hg$<br>[627-44-1]<br>SPIUPAOJDZNUJH-UHFFFAOYSA-N | $1.0\times10^{-3}$ | 3800 | Abraham et al. (2008) | Q | 218 |
| dipropylmercury<br>$C_6H_{14}Hg$<br>[628-85-3]<br>UZTYYBPPVOXULF-UHFFFAOYSA-N | $5.6\times10^{-4}$ | 4600 | Abraham et al. (2008) | Q | 218 |
| diisopropylmercury<br>$C_6H_{14}Hg$<br>[1071-39-2]<br>UVUGOJQWNVFTRT-UHFFFAOYSA-N | $3.9\times10^{-4}$ | 4600 | Abraham et al. (2008) | Q | 218 |
| dibutylmercury<br>$C_8H_{18}Hg$<br>[629-35-6]<br>CCYKQVBIPYDCKS-UHFFFAOYSA-N | $2.9\times10^{-4}$ | 5400 | Abraham et al. (2008) | Q | 218 |
| diphenylmercury<br>$C_{12}H_{10}Hg$<br>[587-85-9]<br>HWMTUNCVVYPZHZ-UHFFFAOYSA-N | $2.8\times10^{2}$ | 8800 | Abraham et al. (2008) | Q | 218 |
| hydroxymethylmercury<br>$CH_3HgOH$<br>[1184-57-2]<br>KRZWEBVPFGCYMY-UHFFFAOYSA-M | $9.8\times10^{2}$<br>$1.5\times10^{3}$ | 7700 | Iverfeldt and Persson (1985)<br>Shon et al. (2005) | M<br>? | <br>823 |
| phenyl mercuric ethanoate<br>$C_8H_8HgO_2$<br>[62-38-4]<br>XEBWQGVWTUSTLN-UHFFFAOYSA-M | $1.5\times10^{4}$ | | Suntio et al. (1988) | V | 12 |
| (3-cyanoguanidino)methylmercury<br>$C_3H_6N_4Hg$<br>(methylmercuric dicyanamide)<br>[502-39-6]<br>JVJUWCMBRUMDDQ-UHFFFAOYSA-N | $7.0\times10^{4}$ | | HSDB (2015) | Q | 99 |



Table A11.9: Mercury (Hg) (. . . continued)

| Substance Formula (Trivial Name) [CAS Registry Number] InChIKey | $H_s^{cp}$ (at $T^\ominus$) $\left[\dfrac{\text{mol}}{\text{m}^3\,\text{Pa}}\right]$ | $\dfrac{\text{d}\ln H_s^{cp}}{\text{d}(1/T)}$ [K] | Reference | Type | Note |
|---|---|---|---|---|---|
| chloromethylmercury $CH_3HgCl$ [115-09-3] BABMCXWQNSQAOC-UHFFFAOYSA-M | $2.2\times10^1$ $1.5\times10^1$ $2.0\times10^1$ $2.6\times10^1$ $1.5\times10^1$ | 1800 4100 5300 | Iverfeldt and Lindqvist (1982) Talmi and Mesmer (1975) WHO (1990) Abraham et al. (2008) Schroeder and Munthe (1998) Iverfeldt and Persson (1985) | M M C Q ? ? | 33, 824 87 33 214 80, 21 219 |
| chloroethylmercury $C_2H_5HgCl$ [107-27-7] QWUGXIXRFGEYBD-UHFFFAOYSA-M | $1.5\times10^1$ | 5600 | Abraham et al. (2008) | Q | 218 |
| chloropropylmercury $C_3H_7HgCl$ [2440-40-6] ZLAYJSKLDWSALK-UHFFFAOYSA-M | $1.2\times10^1$ | 5900 | Abraham et al. (2008) | Q | 218 |
| chloroisopropylmercury $C_3H_7HgCl$ [30615-19-1] YOKZNQICWABELX-UHFFFAOYSA-M | 9.9 | 6000 | Abraham et al. (2008) | Q | 218 |
| chlorobutylmercury $C_4H_9HgCl$ [543-63-5] OKPMTXZRMGMMOO-UHFFFAOYSA-M | 8.8 | 6300 | Abraham et al. (2008) | Q | 218 |
| chloropentylmercury $C_5H_{11}HgCl$ [544-15-0] UHFZINPMKCNPQL-UHFFFAOYSA-M | 7.0 | 6700 | Abraham et al. (2008) | Q | 218 |
| chlorophenylmercury $C_6H_5HgCl$ [100-56-1] AWGTVRDHKJQFAX-UHFFFAOYSA-M | $3.8\times10^2$ $9.2\times10^2$ | 7400 | Abraham et al. (2008) Abraham et al. (2008) | V Q | 218 |
| 2-methoxyethylmercury chloride $CH_3OC_2H_4HgCl$ (aretan) [123-88-6] VJTAZCKMHINUKO-UHFFFAOYSA-M | $3.9\times10^3$ | 8600 | Abraham et al. (2008) | Q | 218 |
| bromomethylmercury $CH_3HgBr$ [506-83-2] ZDHHIJSLJCLMPX-UHFFFAOYSA-M | 3.7 | 4800 | Abraham et al. (2008) Iverfeldt and Persson (1985) | Q ? | 214 219 |



Table A11.9: Mercury (Hg) (...continued)

| Substance<br>Formula<br>(Trivial Name)<br>[CAS Registry Number]<br>InChIKey | $H_s^{cp}$<br>(at $T^\ominus$)<br>$\left[\dfrac{\text{mol}}{\text{m}^3\,\text{Pa}}\right]$ | $\dfrac{\text{d}\ln H_s^{cp}}{\text{d}(1/T)}$<br><br>[K] | Reference | Type | Note |
|---|---|---|---|---|---|
| bromoethylmercury<br>$C_2H_5HgBr$<br>[107-26-6]<br>UREACWLAXSOUKG-UHFFFAOYSA-M | 3.0 | 5200 | Abraham et al. (2008) | Q | 218 |
| bromophenylmercury<br>$C_6H_5HgBr$<br>[1192-89-8]<br>PUPHNPSAIJQNEE-UHFFFAOYSA-M | $1.8\times10^2$ | 6900 | Abraham et al. (2008) | Q | 218 |
| iodomethylmercury<br>$CH_3HgI$<br>[143-36-2]<br>JVDIOYBHEYUIBM-UHFFFAOYSA-M | 2.0<br>$5.8\times10^{-1}$ | 4800 | Abraham et al. (2008)<br>Iverfeldt and Persson (1985) | Q<br>? | 218<br>219 |
| iodoethylmercury<br>$C_2H_5HgI$<br>[2440-42-8]<br>BYIGJUQTUPMNMF-UHFFFAOYSA-M | 2.5 | 5200 | Abraham et al. (2008) | Q | 218 |
| iodophenylmercury<br>$C_6H_5HgI$<br>[823-04-1]<br>BISBXZWWFIOZSX-UHFFFAOYSA-M | $9.0\times10^1$ | 6700 | Abraham et al. (2008) | Q | 218 |





**A11.10   Lead (Pb)**

Table A11.10: Lead (Pb)

| Substance Formula (Trivial Name) [CAS Registry Number] InChIKey | $H_s^{cp}$ (at $T^{\ominus}$) $\left[\dfrac{\text{mol}}{\text{m}^3\,\text{Pa}}\right]$ | $\dfrac{\text{d}\ln H_s^{cp}}{\text{d}(1/T)}$ [K] | Reference | Type | Note |
|---|---|---|---|---|---|
| tetramethyl lead C$_4$H$_{12}$Pb [75-74-1] XOOGZRUBTYCLHG-UHFFFAOYSA-N | $1.6\times10^{-5}$ | | HSDB (2015) | V | |
| ethyltrimethylplumbane C$_5$H$_{14}$Pb [1762-26-1] KHQJREYATBQBHY-UHFFFAOYSA-N | $2.8\times10^{-5}$ | | HSDB (2015) | Q | 99 |
| diethyldimethylplumbane C$_6$H$_{16}$Pb (diethyldimethyl lead) [1762-27-2] OLOAJSHVLXNSQV-UHFFFAOYSA-N | $2.1\times10^{-5}$ | | HSDB (2015) | Q | 99 |
| triethylmethylplumbane C$_7$H$_{18}$Pb (methyltriethyl lead) [1762-28-3] KGFRUGHBHNUHOS-UHFFFAOYSA-N | $1.6\times10^{-5}$ | | HSDB (2015) | Q | 99 |
| tetraethyl lead C$_8$H$_{20}$Pb [78-00-2] MRMOZBOQVYRSEM-UHFFFAOYSA-N | $1.3\times10^{-5}$ $1.3\times10^{-5}$ | 6400 | Feldhake and Stevens (1963) Abraham (1979) | M ? | |
| trimethyl lead chloride C$_3$H$_9$ClPb [1520-78-1] HPQRSQFZILKRDH-UHFFFAOYSA-M | 2.5 | | Ebert et al. (2023) | ? | 316 |
| triethyl lead chloride C$_6$H$_{15}$ClPb [1067-14-7] UQWYUMLFPUILRT-UHFFFAOYSA-M | $2.1\times10^{5}$ $1.1\times10^{-3}$ | | Duchowicz et al. (2020) Duchowicz et al. (2020) | V Q | 186 |



**Appendix: Notes**

**1)** A detailed temperature dependence with more than one parameter is available in the original publication. Here, only the temperature dependence at 298.15 K according to the van 't Hoff equation is presented.

**2)** Clever et al. (2014) recommend the data from Rettich et al. (2000).

**3)** Vapor pressure data for water from Wagner and Pruss (1993) was needed to calculate $H_s$.

**4)** The data from Millero et al. (2002a) were fitted to the 3-parameter equation: $H_s^{cp} = \exp(-130.91491 + 6700.12242/T + 17.04684 \ln(T))$ mol/(m$^3$ Pa) with $T$ in K.

**5)** Almost the same data were also published in Millero et al. (2002b).

**6)** The data from Millero et al. (2002b) were fitted to the 3-parameter equation: $H_s^{cp} = \exp(-118.73105 + 6163.97787/T + 15.22401 \ln(T))$ mol/(m$^3$ Pa) with $T$ in K.

**7)** Almost the same data were also published in Millero et al. (2002a).

**8)** The data from Rettich et al. (2000) were fitted to the 3-parameter equation: $H_s^{cp} = \exp(-179.13831 + 8707.17767/T + 24.33473 \ln(T))$ mol/(m$^3$ Pa) with $T$ in K.

**9)** The data from Sherwood et al. (1991) were fitted to the 3-parameter equation: $H_s^{cp} = \exp(-197.67462 + 9515.09306/T + 27.11204 \ln(T))$ mol/(m$^3$ Pa) with $T$ in K.

**10)** The data from Rettich et al. (1981) were fitted to the 3-parameter equation: $H_s^{cp} = \exp(-178.21340 + 8672.23354/T + 24.19307 \ln(T))$ mol/(m$^3$ Pa) with $T$ in K.

**11)** Measured at high temperature and extrapolated to $T^{\ominus} = 298.15$ K.

**12)** Value at $T = 293$ K.

**13)** Value at $T = 273$ K.

**14)** Value at $T = 310$ K.

**15)** The data from Murray and Riley (1969) were fitted to the 3-parameter equation: $H_s^{cp} = \exp(-180.22078 + 8760.50130/T + 24.49289 \ln(T))$ mol/(m$^3$ Pa) with $T$ in K.

**16)** The data from Shoor et al. (1969) were fitted to the 3-parameter equation: $H_s^{cp} = \exp(-91.44799 + 4548.67245/T + 11.38821 \ln(T))$ mol/(m$^3$ Pa) with $T$ in K.

**17)** The data from Carpenter (1966) were fitted to the 3-parameter equation: $H_s^{cp} = \exp(-130.04464 + 6687.45227/T + 16.90114 \ln(T))$ mol/(m$^3$ Pa) with $T$ in K.

**18)** The data from Morrison and Billett (1952) were fitted to the 3-parameter equation: $H_s^{cp} = \exp(-167.89288 + 8254.02144/T + 22.62741 \ln(T))$ mol/(m$^3$ Pa) with $T$ in K.

**19)** The data from Winkler (1891b) were fitted to the 3-parameter equation: $H_s^{cp} = \exp(-155.30315 + 7638.78869/T + 20.77945 \ln(T))$ mol/(m$^3$ Pa) with $T$ in K.

**20)** Calculated using machine learning matrix completion methods (MCMs).

**21)** Several references are given in the list of Henry's law constants but not assigned to specific species.

**22)** The data from Dean and Lange (1999) were fitted to the 3-parameter equation: $H_s^{cp} = \exp(-161.84252 + 7966.66767/T + 21.73409 \ln(T))$ mol/(m$^3$ Pa) with $T$ in K.

**23)** The partial pressure of water vapor (needed to convert some Henry's law constants) was calculated using the formula given by Buck (1981). The quantities $A$ and $\alpha$ from Dean and Lange (1999) were assumed to be identical.

**24)** Value at "room temperature".

**25)** Clever et al. (2014) recommend the data from Battino (1981).

**26)** Battino (1981) concludes that ozone aqueous chemistry needs further clarification. Data from Roth and Sullivan (1981) are recommended, in spite of limitations and assumptions of the data.

**27)** Roth and Sullivan (1981) found that $H_s$ depends on the concentration of OH$^-$.

**28)** Value at $T = 291$ K.

**29)** Value given here as quoted by Durham et al. (1981).

**30)** Lide and Frederikse (1995) present an unusually low value for the Henry solubility of ozone. They refer to Battino (1981) as the source but the quoted value cannot be found there.

**31)** Parker (1992) assume that the free energy of solvation of atomic hydrogen is equal to that of He because of a similar van der Waals radius.

**32)** Roduner and Bartels (1992) say that the free energy of solvation $\Delta G_{\mathrm{solv}}^{\mathrm{H}}$ (and therefore Henry's law constant) of atomic hydrogen is approximated well by that of molecular hydrogen. However, they apparently do not give a value for $\Delta G_{\mathrm{solv}}^{\mathrm{H}}$.

**33)** Fitting the temperature dependence $\mathrm{d}\ln H/\mathrm{d}(1/T)$ produced a low correlation coefficient ($r^2 < 0.9$). The data should be treated with caution.

**34)** Data digitized from Figs. 2 and 3. in Schmidt (1979).



35) The data from Gordon et al. (1977) were fitted to the 3-parameter equation: $H_s^{cp} = \exp(-158.95051 + 6959.76267/T + 21.73478\ln(T))$ mol/(m$^3$ Pa) with $T$ in K.

36) The data from Crozier and Yamamoto (1974) were fitted to the 3-parameter equation: $H_s^{cp} = \exp(-129.44163 + 5676.58091/T + 17.31002\ln(T))$ mol/(m$^3$ Pa) with $T$ in K.

37) The data presented for hydrogen in Tab. II of Shoor et al. (1969) appear to be incorrect and are not reproduced here.

38) Value at $T$ = 303 K.

39) The data from Morrison and Billett (1952) were fitted to the 3-parameter equation: $H_s^{cp} = \exp(-94.36490 + 4110.23880/T + 12.07743\ln(T))$ mol/(m$^3$ Pa) with $T$ in K.

40) The data from Braun (1900) were fitted to the 3-parameter equation: $H_s^{cp} = \exp(171.59451 - 6856.02728/T - 28.14739\ln(T))$ mol/(m$^3$ Pa) with $T$ in K.

41) The data from Winkler (1891a) were fitted to the 3-parameter equation: $H_s^{cp} = \exp(-103.47250 + 4506.63123/T + 13.44160\ln(T))$ mol/(m$^3$ Pa) with $T$ in K.

42) Fitting the temperature dependence $d\ln H/d(1/T)$ produced a very low correlation coefficient ($r^2 < 0.5$). The data should be treated with caution.

43) The paper by Bunsen (1855a) was written in German. English versions with the same data were published by Bunsen (1855b) and Bunsen (1855c).

44) The data from Dean and Lange (1999) were fitted to the 3-parameter equation: $H_s^{cp} = \exp(-98.78036 + 4298.15060/T + 12.74131\ln(T))$ mol/(m$^3$ Pa) with $T$ in K.

45) Young (1981a) recommend the data from Muccitelli and Wen (1978).

46) The data from Muccitelli and Wen (1978) were fitted to the 3-parameter equation: $H_s^{cp} = \exp(-103.11330 + 4676.56978/T + 13.28348\ln(T))$ mol/(m$^3$ Pa) with $T$ in K.

47) Calculated from correlation between the polarizabilities and solubilities of stable gases. The temperature dependence is an estimate of the upper limit.

48) Jacob (1986) assumed the temperature dependence to be the same as for water.

49) In the abstract, Schwartz (1984) gives a range of 9.9 mol/(m$^3$ Pa) $< H_s^{cp} < 3.0{\times}10^1$ mol/(m$^3$ Pa). The mean value of this range ($2.0{\times}10^1$ mol/(m$^3$ Pa)) has been used by Lelieveld and Crutzen (1991), Pandis and Seinfeld (1989), and Jacob (1986).

50) The value of $H_s^{\ominus}$ was taken from Schwartz (1984).

51) Erratum for page 264 of Fogg and Sangster (2003): The second value from their Ref. [10] refers to 291.15 K, not 281.15 K.

52) This value is a correction of the solubility published by Lind and Kok (1986).

53) This value was measured at low pH. It is superseded by a later publication of the same group (Lind and Kok, 1994).

54) Pandis and Seinfeld (1989) cite an incorrect value from Lind and Kok (1986), see erratum by Lind and Kok (1994).

55) The data from Rettich et al. (1984) were fitted to the 3-parameter equation: $H_s^{cp} = \exp(-187.67954 + 8903.42524/T + 25.60079\ln(T))$ mol/(m$^3$ Pa) with $T$ in K.

56) The data from Murray et al. (1969) were fitted to the 3-parameter equation: $H_s^{cp} = \exp(-174.95275 + 8370.22025/T + 23.67878\ln(T))$ mol/(m$^3$ Pa) with $T$ in K.

57) The data from Morrison and Billett (1952) were fitted to the 3-parameter equation: $H_s^{cp} = \exp(-193.68175 + 9249.63150/T + 26.45117\ln(T))$ mol/(m$^3$ Pa) with $T$ in K.

58) Value at $T$ = 311 K.

59) The data from Braun (1900) were fitted to the 3-parameter equation: $H_s^{cp} = \exp(291.66324 - 11637.66767/T - 46.44134\ln(T))$ mol/(m$^3$ Pa) with $T$ in K.

60) The data from Winkler (1891b) were fitted to the 3-parameter equation: $H_s^{cp} = \exp(-164.15156 + 7906.86704/T + 22.05399\ln(T))$ mol/(m$^3$ Pa) with $T$ in K.

61) The data from Dean and Lange (1999) were fitted to the 3-parameter equation: $H_s^{cp} = \exp(-163.64571 + 7887.30480/T + 21.97696\ln(T))$ mol/(m$^3$ Pa) with $T$ in K.

62) Tsuji et al. (1990) provide effective Henry's law constants at several pH values. Here, only the value at pH 5.8 is shown for the (acidic) S-compounds, and the value at pH 8.6 for the alkaline N-compounds.

63) Value given here as quoted by Betterton (1992).

64) The data from Dean and Lange (1999) were fitted to the 3-parameter equation: $H_s^{cp} = \exp(206.08500 - 7165.18642/T - 32.18383\ln(T))$ mol/(m$^3$ Pa) with $T$ in K.

65) Bone et al. (1983) gives Carter et al. (1968) as the source. However, no data were found in that reference.

66) There is a typo in Sander et al. (2011): The value for $A$ should be $-10.19$, not $10.19$.





67) Modarresi et al. (2007) use different descriptors for their calculations. They conclude that GA/RBFN is the best QSPR model. Only these results are shown here.

68) Incorrect data are given by Burkholder et al. (2019) for HN$_3$. The correct parameter for the temperature dependence is: A = -10.19 (R. E. Huie, pers. comm. 2021).

69) Incorrect data are given by Burkholder et al. (2015) for HN$_3$. The correct parameter for the temperature dependence is: A = -10.19 (R. E. Huie, pers. comm. 2021).

70) Solubility in sea water.

71) The data from Weiss and Price (1980) were fitted to the 3-parameter equation: $H_s^{cp} = \exp(-180.63611 + 9824.20147/T + 24.46112\ln(T))$ mol/(m$^3$ Pa) with $T$ in K.

72) Value at $T = 296$ K.

73) The data from Roth (1897) were fitted to the 3-parameter equation: $H_s^{cp} = \exp(-125.17909 + 7706.80638/T + 15.96486\ln(T))$ mol/(m$^3$ Pa) with $T$ in K.

74) Value given here as quoted by Gabel and Schultz (1973).

75) Value given here as quoted by Sy and Hasbrouck (1964).

76) The H298 and A,B,C data listed in Table 5.4 of Sander et al. (2011) are inconsistent, with 94 % difference.

77) The H298 and A,B,C data listed in Table 5.4 of Sander et al. (2006) are inconsistent, with 94 % difference.

78) A minus sign is missing in the fitting parameter presented by Young (1981b). It should be -62.8086, not 62.8086.

79) Value at $T = 297$ K.

80) Value at $T = 288$ K.

81) The data from Winkler (1901) were fitted to the 3-parameter equation: $H_s^{cp} = \exp(-184.00012 + 8924.34832/T + 25.13228\ln(T))$ mol/(m$^3$ Pa) with $T$ in K.

82) The data from Loomis (1928) were fitted to the 3-parameter equation: $H_s^{cp} = \exp(-223.88313 + 10620.37030/T + 31.13453\ln(T))$ mol/(m$^3$ Pa) with $T$ in K.

83) The data from Dean and Lange (1999) were fitted to the 3-parameter equation: $H_s^{cp} = \exp(-160.19223 + 7888.02642/T + 21.56401\ln(T))$ mol/(m$^3$ Pa) with $T$ in K.

84) Incorrect data are given by Burkholder et al. (2019) for NO. The correct parameters for the temperature dependence are: A = -163.86, B = 8234, C = 22.816 (R. E. Huie, pers. comm. 2021).

85) Incorrect data are given by Burkholder et al. (2015) for NO. The correct parameters for the temperature dependence are: A = -163.86, B = 8234, C = 22.816 (R. E. Huie, pers. comm. 2021).

86) The fitting parameters $A$, $B$, $C$, and $D$ in Table I of Wilhelm et al. (1977) do not reproduce the data in their Table III.

87) Value at $T = 295$ K.

88) Pandis and Seinfeld (1989) refer to Schwartz (1984) as the source but the quoted value cannot be found there.

89) Value obtained by estimating the diffusion coefficient for NO$_3$ to be $D = 1.0\times10^{-5}$ cm$^2$/s.

90) Jacob (1986) assume that NO$_3$ has the same Henry's law constant as HNO$_3$.

91) Seinfeld and Pandis (1998) probably refer to the incorrect value given by Pandis and Seinfeld (1989).

92) Calculated from the solvation free energy.

93) Calculated from the solvation free energy.

94) Calculated from the solvation free energy.

95) This value was extrapolated from data at $T = 230$ K and $T = 273$ K.

96) Robinson et al. (1997) applied an empirical correlation between Henry's law solubilities and boiling points from Schwartz and White (1981).

97) Estimate based on the relation between boiling points and Henry's law constants for other nitrogen oxides from Schwartz and White (1981).

98) Fast, irreversible hydrolysis is assumed, which is equivalent to an infinite effective Henry's law constant.

99) Calculated based on the method by Meylan and Howard (1991).

100) Lelieveld and Crutzen (1991) assume the temperature dependence to be the same as for $a(\mathrm{H^+})a(\mathrm{NO_3^-})/p(\mathrm{HNO_3})$ in Schwartz and White (1981).

101) $H_s' = 2.6 \times 10^7 \times \exp\left(8700\,\mathrm{K}\left(\dfrac{1}{T} - \dfrac{1}{T^\ominus}\right)\right)\dfrac{\mathrm{mol}^2}{\mathrm{m^6\,Pa}}$

102) $H_s' = 2.4 \times 10^7 \times \exp\left(8700\,\mathrm{K}\left(\dfrac{1}{T} - \dfrac{1}{T^\ominus}\right)\right)\dfrac{\mathrm{mol}^2}{\mathrm{m^6\,Pa}}$

103) Pandis and Seinfeld (1989) refer to Schwartz (1984) as the source but it is probably from Schwartz and White (1981).

104) The value is incorrect. See erratum by Brimblecombe and Clegg (1989).

105) Möller and Mauersberger (1992) assumed the solubility of HNO$_4$ to be comparable to that of HNO$_3$.

106) $H_s' = 9.4 \times 10^1 \times \exp\left(7400\,\mathrm{K}\left(\dfrac{1}{T} - \dfrac{1}{T^\ominus}\right)\right)\dfrac{\mathrm{mol}^2}{\mathrm{m^6\,Pa}}$



**107)** The data from Dean et al. (1973) were fitted to the 3-parameter equation: $H_s^{cp} = \exp(-164.52717 + 8214.77776/T + 21.97482\ln(T))$ mol/(m$^3$ Pa) with $T$ in K.

**108)** The data from Ashton et al. (1968) were fitted to the 3-parameter equation: $H_s^{cp} = \exp(-279.21972 + 13536.60588/T + 38.97386\ln(T))$ mol/(m$^3$ Pa) with $T$ in K.

**109)** The data from Dean et al. (1973) were fitted to the 3-parameter equation: $H_s^{cp} = \exp(-318.29953 + 15733.17858/T + 44.55320\ln(T))$ mol/(m$^3$ Pa) with $T$ in K.

**110)** The value presented here appears to be the total solubility of chlorine (i.e., the sum of $Cl_2$ and HOCl) at a partial pressure of $p(Cl_2) = 101325$ Pa. This is different from Henry's law constant, which is defined at extrapolation to infinite dilution.

**111)** Young (1983) recommends values calculated from Table 1 of Adams and Edmonds (1937). Thus, the data refer to effective values that take into account the hydrolysis in the aqueous-phase:

$$H_{s,eff} = ([Cl_2] + [HOCl])/p(Cl_2)$$

In addition, the values apply to a partial pressure of $p(Cl_2) = 101325$ Pa, and not to infinite dilution.

**112)** The same experimental data were also published by Whitney and Vivian (1941b).

**113)** The data from Yakovkin (1900) were fitted to the 3-parameter equation: $H_s^{cp} = \exp(-122.31264 + 7690.40834/T + 15.63947\ln(T))$ mol/(m$^3$ Pa) with $T$ in K.

**114)** Leist (1986) converted the total solubility of chlorine in pure water from Adams and Edmonds (1937) to an intrinsic Henry's law constant.

**115)** Adams and Edmonds (1937) re-analyzed the data from Yakovkin (1900) and Arkadiev (1918), considering deviations from the perfect gas law. They calculated the total solubility of chlorine (i.e., the sum of $Cl_2$ and HOCl) at several partial pressures of $Cl_2$. This is different from Henry's law constant, which is defined at extrapolation to infinite dilution.

**116)** Arkadiev (1918) re-analyzed the measurements of Yakovkin (1900). In addition to the data between 15 °C and 83.4 °C, he also analyzed the experimental results at 0 °C and obtained a dimensionless Henry solubility of $H_s^{cc} = 4.115$ at that temperature.

**117)** The value of $\Delta H^\circ$ listed in Table 2 of Bartlett and Margerum (1999) is incorrect. The correct value can be found in the text on page 3411.

**118)** Wilhelm et al. (1977) present a fitting function for $Cl_2$ based on four papers which are cited in the footnotes of Table I. However, Bunsen (1855b) and Bunsen (1855c) don't contain any data for $Cl_2$, and the data from Whitney and Vivian (1941a) and Whitney and Vivian (1941b) are inconsistent with the fitting function.

**119)** Calculated from the free energy of solution by Schwarz and Dodson (1984).

**120)** $H_s' = 2.0\times10^7 \ \dfrac{\text{mol}^2}{\text{m}^6\,\text{Pa}}$

**121)** $H_s' = 2.0\times10^7 \times \exp\left(9000\ \text{K}\left(\dfrac{1}{T} - \dfrac{1}{T^\ominus}\right)\right)\dfrac{\text{mol}^2}{\text{m}^6\,\text{Pa}}$

**122)** $H_s' = 2.0\times10^7 \times \exp\left(9000\ \text{K}\left(\dfrac{1}{T} - \dfrac{1}{T^\ominus}\right)\right)\dfrac{\text{mol}^2}{\text{m}^6\,\text{Pa}}$

**123)** $H_s' = 2.0\times10^7 \times \exp\left(9000\ \text{K}\left(\dfrac{1}{T} - \dfrac{1}{T^\ominus}\right)\right)\dfrac{\text{mol}^2}{\text{m}^6\,\text{Pa}}$

**124)** The data from Dean and Lange (1999) were fitted to the 3-parameter equation: $H_s^{cp} = \exp(9.16427 + 45.13997/T - 1.92853\ln(T))$ mol/(m$^3$ Pa) with $T$ in K.

**125)** Pandis and Seinfeld (1989) refer to Marsh and McElroy (1985) as the source but the quoted value cannot be found there.

**126)** This value was extrapolated from data at $T = 215$ K and $T = 263$ K.

**127)** Value at pH = 6.5.

**128)** Value at $T = 200$ K.

**129)** Secoy and Cady (1941) measured the gas/aqueous equilibrium constant $p(Cl_2O)/c(HOCl)^2$ but not the intrinsic Henry's law constant of $Cl_2O$.

**130)** Ourisson and Kastner (1939) measured the gas/aqueous equilibrium constant $p(Cl_2O)/c(HOCl)^2$ but not the intrinsic Henry's law constant of $Cl_2O$.

**131)** The data from this work were fitted to the 3-parameter equation: $H_s^{cp} = \exp(1680.49677 - 69933.08019/T - 254.37188\ln(T))$ mol/(m$^3$ Pa) with $T$ in K.

**132)** The gas/aqueous equilibrium constant $p(Cl_2O)/c(HOCl)^2$ was combined with the temperature-dependent aqueous-phase hydration constant $c(HOCl)^2/c(Cl_2O)$ from Roth (1929) in order to calculate the intrinsic Henry's law constant of $Cl_2O$.

**133)** Data for the equilibrium between gaseous $Cl_2O$ and aqueous HOCl were taken from Secoy and Cady (1941).

**134)** Data for the equilibrium between gaseous $Cl_2O$ and aqueous HOCl were taken from Ourisson and Kastner (1939).

**135)** Value at $T = 277$ K.



**136)** The recommended value from Wilhelm et al. (1977) appears to be dubious as it refers to Secoy and Cady (1941) who do not provide a value for the intrinsic Henry's law constant of $Cl_2O$.

**137)** Young (1983) cites data from Secoy and Cady (1941). However, that paper only describes the equilibrium between gas-phase $Cl_2O$ and aqueous-phase HOCl. A Henry's law constant of $Cl_2O$ is not provided. In addition, the values given by Young (1983) are not extrapolated to infinite dilution but to 1 atm partial pressure of $Cl_2O$. It is not explained how the nonlinear pressure dependence was extrapolated to 1 atm.

**138)** Wilhelm et al. (1977) cite Secoy and Cady (1941) as the source for their value. However, that paper only describes the equilibrium between gas-phase $Cl_2O$ and aqueous-phase HOCl. A Henry's law constant of $Cl_2O$ is not provided.

**139)** Even though Haller and Northgraves (1955) have been cited several times as the source of the $ClO_2$ solubility data, they did not perform any measurements. They took the data from the 1952 edition of the Kirk-Othmer Encyclopedia of Chemical Technology which apparently reproduced data from Holst (1944).

**140)** Derived as a fitting parameter used in numerical modeling.

**141)** Robinson et al. (1997) assumed that the entropy of vaporization is the same for HOCl and $ClNO_3$ according to Trouton's rule. On their page 3592, they mention a value of 7 M/atm at 250 K. However, checking their Fig. 9 and applying the temperature-dependence equation from their Tab. 3, it seems that the value of 7 M/atm refers to 298 K, not 250 K.

**142)** Dubik et al. (1987) measured the solubility in concentrated salt solutions (natural brines).

**143)** Value given here as quoted by McCoy et al. (1990).

**144)** The data from Dean and Lange (1999) were fitted to the 3-parameter equation: $H_s^{cp} = \exp(-148.75612 + 9709.79389/T + 19.53402 \ln(T))$ $mol/(m^3\,Pa)$ with $T$ in K.

**145)** $H_s' = 8.2 \times 10^9 \times \exp\left(10000\,\mathrm{K}\left(\frac{1}{T} - \frac{1}{T^{\ominus}}\right)\right) \frac{mol^2}{m^6\,Pa}$

**146)** $H_s' = 1.3 \times 10^{10} \times \exp\left(10000\,\mathrm{K}\left(\frac{1}{T} - \frac{1}{T^{\ominus}}\right)\right) \frac{mol^2}{m^6\,Pa}$

**147)** $H_s' = 7.0 \times 10^9 \times \exp\left(10000\,\mathrm{K}\left(\frac{1}{T} - \frac{1}{T^{\ominus}}\right)\right) \frac{mol^2}{m^6\,Pa}$

**148)** Chameides and Stelson (1992) give a value of $H_s' = 7.1 \times 10^9 \times \exp\left(6100\,\mathrm{K}\left(\frac{1}{T} - \frac{1}{T^{\ominus}}\right)\right) \frac{mol^2}{m^6\,Pa}$. They refer to Jacob (1986) and Chameides (1984) as the source but this value cannot be found there.

**149)** The data from Dean and Lange (1999) were fitted to the 3-parameter equation: $H_s^{cp} = \exp(-60.28318 + 2830.41867/T + 8.66642 \ln(T))$ $mol/(m^3\,Pa)$ with $T$ in K.

**150)** The value is from Table 1 of the paper. However, *J. Geophys. Res.* forgot to print the tables and I received them directly from the author.

**151)** The value presented for HOBr is incorrect. A corrected version was later published by Burkholder et al. (2019).

**152)** Fickert (1998) extracted a value from wetted-wall flow tube experiments. However, it was later discovered that under the experimental conditions no evaluation of $H_s$ is possible (J. Crowley, pers. comm., 1999).

**153)** Value at $T = 275$ K.

**154)** Value at $T = 290$ K.

**155)** Calculated using data from Wagman et al. (1982) and the aqueous-phase equilibrium $Cl_2 + Br_2 \rightleftharpoons 2\,BrCl$ from Wang et al. (1994).

**156)** Thompson and Zafiriou (1983) quote a paper as the source that gives only the solubility but not the Henry's law constant.

**157)** Calculated from the free energy of solution by Schwarz and Bielski (1986).

**158)** $H_s' = 2.5 \times 10^{10} \times \exp\left(9800\,\mathrm{K}\left(\frac{1}{T} - \frac{1}{T^{\ominus}}\right)\right) \frac{mol^2}{m^6\,Pa}$

**159)** $H_s' = 2.1 \times 10^{10} \times \exp\left(9800\,\mathrm{K}\left(\frac{1}{T} - \frac{1}{T^{\ominus}}\right)\right) \frac{mol^2}{m^6\,Pa}$

**160)** Saiz-Lopez et al. (2014) refer to Saiz-Lopez et al. (2008) as the source but the quoted value cannot be found there.

**161)** It is unclear to which isomer the value of the Henry's law constant refers to.

**162)** Assumed to be infinity by analogy with $INO_3$.

**163)** Thompson and Zafiriou (1983) assume that $H_s^{cp}(HOI)$ is between $4.4 \times 10^{-1}\,mol/(m^3\,Pa)$ and $4.4 \times 10^2\,mol/(m^3\,Pa)$.

**164)** Badia et al. (2019) assume that $INO_2$ has the same Henry's law constant as $BrNO_2$.

**165)** Data taken from the AGRITOX database file agritox-20210608.zip.



**166)** Fogg and Young (1988) provide two different fitting functions: One for $T < 283.2$ K and one for $T > 283.2$ K. At $T = 283.2$ K, the functions have different values and different slopes. Here, only the function that is valid at $T^\ominus$ is used.

**167)** The data from Clarke and Glew (1971) were fitted to the 3-parameter equation: $H_s^{cp} = \exp(-133.37135 + 7422.07576/T + 17.82903 \ln(T))$ mol/(m³ Pa) with $T$ in K.

**168)** The data from Schoenfeld (1855) were fitted to the 3-parameter equation: $H_s^{cp} = \exp(98.96644 - 3021.28876/T - 16.78233 \ln(T))$ mol/(m³ Pa) with $T$ in K.

**169)** The data from Dean and Lange (1999) were fitted to the 3-parameter equation: $H_s^{cp} = \exp(-122.57010 + 6962.28299/T + 16.20245 \ln(T))$ mol/(m³ Pa) with $T$ in K.

**170)** The parameter fit for the temperature dependence is incorrect. A corrected version was later presented by Iliuta and Larachi (2007).

**171)** The data from Clarke and Glew (1971) were fitted to the 3-parameter equation: $H_s^{cp} = \exp(-152.96053 + 8324.82999/T + 20.73129 \ln(T))$ mol/(m³ Pa) with $T$ in K.

**172)** Obtained with $D_2O$ as solvent.

**173)** Value at $T = 353$ K.

**174)** The data from Schoenfeld (1855) were fitted to the 3-parameter equation: $H_s^{cp} = \exp(265.79241 - 9131.99684/T - 42.01987 \ln(T))$ mol/(m³ Pa) with $T$ in K.

**175)** Value given here as quoted by Rodríguez-Sevilla et al. (2002).

**176)** The data from Dean and Lange (1999) were fitted to the 3-parameter equation: $H_s^{cp} = \exp(153.05871 - 4328.05304/T - 25.05397 \ln(T))$ mol/(m³ Pa) with $T$ in K.

**177)** Marti et al. (1997) give partial pressures of $H_2SO_4$ over a concentrated solution (e.g., $2.6 \times 10^{-9}$ Pa for 54.1 weight-percent at 298 K). Extrapolating this to dilute solutions can only be considered an order-of-magnitude approximation for $H_s$.

**178)** Ayers et al. (1980) give partial pressures of $H_2SO_4$ over concentrated solutions at high temperatures. Extrapolating this to dilute solutions can only be considered an order-of-magnitude approximation for $H_s$.

**179)** Gmitro and Vermeulen (1964) give partial pressures of $H_2SO_4$ over a concentrated solution (e.g., $10^{-7}$ mmHg for 70 weight-percent at 298 K). Extrapolating this to dilute solutions can only be

considered an order-of-magnitude approximation for $H_s$.

**180)** Clegg et al. (1998) estimate a Henry's law constant of $5 \times 10^{11}$ atm⁻¹ at 303.15 K for the reaction $H_2SO_4(g) \rightleftharpoons 2\,H^+(aq) + SO_4^{2-}(aq)$ but don't give a definition for it. Probably it is defined as $x^2(H^+) \times x(SO_4^{2-})/p(H_2SO_4)$, where $x$ is the aqueous-phase mixing ratio.

**181)** The data from Bullister et al. (2002) were fitted to the 3-parameter equation: $H_s^{cp} = \exp(-281.50843 + 14256.43847/T + 38.73689 \ln(T))$ mol/(m³ Pa) with $T$ in K.

**182)** The data presented for SF6 in Tab. II of Shoor et al. (1969) appear to be incorrect and are not reproduced here.

**183)** The data from Ashton et al. (1968) were fitted to the 3-parameter equation: $H_s^{cp} = \exp(-431.90650 + 20715.81650/T + 61.33841 \ln(T))$ mol/(m³ Pa) with $T$ in K.

**184)** Value from the validation set for checking whether the model is satisfactory for compounds that are absent from the training set.

**185)** Experimental value, extracted from HENRYWIN.

**186)** Estimation based on the quotient between vapor pressure and water solubility, extracted from HENRYWIN.

**187)** The data presented for helium in Tab. II of Shoor et al. (1969) appear to be incorrect and are not reproduced here.

**188)** The data from Morrison and Johnstone (1954) were fitted to the 3-parameter equation: $H_s^{cp} = \exp(-267.15298 + 11440.04263/T + 37.95994 \ln(T))$ mol/(m³ Pa) with $T$ in K.

**189)** The data from Lannung (1930) were fitted to the 3-parameter equation: $H_s^{cp} = \exp(84.35043 - 4135.59197/T - 14.55881 \ln(T))$ mol/(m³ Pa) with $T$ in K.

**190)** Calculated employing molecular force field models for the solutes from Warr et al. (2015).

**191)** The data from Dean and Lange (1999) were fitted to the 3-parameter equation: $H_s^{cp} = \exp(-153.15219 + 6434.36008/T + 20.89911 \ln(T))$ mol/(m³ Pa) with $T$ in K.

**192)** The data from Morrison and Johnstone (1954) were fitted to the 3-parameter equation: $H_s^{cp} = \exp(-171.84866 + 7492.61303/T + 23.58966 \ln(T))$ mol/(m³ Pa) with $T$ in K.

**193)** The data from Lannung (1930) were fitted to the 3-parameter equation: $H_s^{cp} = \exp(-40.04033 + 1266.80589/T + 4.12574 \ln(T))$ mol/(m³ Pa) with $T$ in K.



**194)** The data from Dean and Lange (1999) were fitted to the 3-parameter equation: $H_s^{cp} = \exp(-150.94728 + 6639.96438/T + 20.42365 \ln(T))$ mol/(m³ Pa) with $T$ in K.

**195)** The data from Rettich et al. (1992) were fitted to the 3-parameter equation: $H_s^{cp} = \exp(-178.55165 + 8674.63293/T + 24.26764 \ln(T))$ mol/(m³ Pa) with $T$ in K.

**196)** The data from Murray and Riley (1970) were fitted to the 3-parameter equation: $H_s^{cp} = \exp(-151.84230 + 7548.13106/T + 20.24085 \ln(T))$ mol/(m³ Pa) with $T$ in K.

**197)** The data from Shoor et al. (1969) were fitted to the 3-parameter equation: $H_s^{cp} = \exp(-177.19900 + 8740.49327/T + 23.99118 \ln(T))$ mol/(m³ Pa) with $T$ in K.

**198)** The data from Ashton et al. (1968) were fitted to the 3-parameter equation: $H_s^{cp} = \exp(-160.52023 + 7898.05096/T + 21.56102 \ln(T))$ mol/(m³ Pa) with $T$ in K.

**199)** The data from Morrison and Johnstone (1954) were fitted to the 3-parameter equation: $H_s^{cp} = \exp(-159.49603 + 7859.86242/T + 21.39868 \ln(T))$ mol/(m³ Pa) with $T$ in K.

**200)** The data from Lannung (1930) were fitted to the 3-parameter equation: $H_s^{cp} = \exp(-183.19260 + 8856.79081/T + 24.97248 \ln(T))$ mol/(m³ Pa) with $T$ in K.

**201)** Calculated employing molecular force field models for the solutes from Vrabec et al. (2001).

**202)** The data from Dean and Lange (1999) were fitted to the 3-parameter equation: $H_s^{cp} = \exp(-143.77232 + 7158.59719/T + 19.05403 \ln(T))$ mol/(m³ Pa) with $T$ in K.

**203)** The data from Morrison and Johnstone (1954) were fitted to the 3-parameter equation: $H_s^{cp} = \exp(-153.87925 + 7855.39037/T + 20.51280 \ln(T))$ mol/(m³ Pa) with $T$ in K.

**204)** Two series of measurements with considerably different results are presented by von Antropoff (1910) for krypton.

**205)** The data from Dean and Lange (1999) were fitted to the 3-parameter equation: $H_s^{cp} = \exp(-220.92114 + 10903.79433/T + 30.49407 \ln(T))$ mol/(m³ Pa) with $T$ in K.

**206)** The value $b$ for Xe given by Himmelblau (1960) in their Table III is incorrect. Most likely, only a minus sign is missing.

**207)** The data from Morrison and Johnstone (1954) were fitted to the 3-parameter equation: $H_s^{cp} = \exp(-165.83721 + 8808.62019/T + 22.15186 \ln(T))$ mol/(m³ Pa) with $T$ in K.

**208)** The data from Dean and Lange (1999) were fitted to the 3-parameter equation: $H_s^{cp} = \exp(-199.40126 + 10306.10786/T + 27.18844 \ln(T))$ mol/(m³ Pa) with $T$ in K.

**209)** The data from Lewis et al. (1987) were fitted to the 3-parameter equation: $H_s^{cp} = \exp(5.03587 + 1555.37916/T - 3.42648 \ln(T))$ mol/(m³ Pa) with $T$ in K.

**210)** Calculated employing molecular force field models for the solutes from Mick et al. (2016).

**211)** The data from Dean and Lange (1999) were fitted to the 3-parameter equation: $H_s^{cp} = \exp(-240.66156 + 12686.97685/T + 33.12171 \ln(T))$ mol/(m³ Pa) with $T$ in K.

**212)** The data from Sisi et al. (1971) were fitted to the 3-parameter equation: $H_s^{cp} = \exp(-81.82525 + 4954.57763/T + 10.19950 \ln(T))$ mol/(m³ Pa) with $T$ in K.

**213)** Solubility in natural sea water. Measurements at different salinities were also performed, but only at a fixed temperature of 296.15 K.

**214)** Temperature dependence calculated using linear free energy relationships (LFER).

**215)** Petersen et al. (1998) give the invalid unit "mol L⁻¹ ppm⁻¹". Here, it is assumed that "ppm" is used as a synonym for "$10^{-6}$ atm".

**216)** Shon et al. (2005) refer to Petersen et al. (1998) as the source but a different value is listed there.

**217)** Value at $T = 333$ K.

**218)** Calculated using linear free energy relationships (LFER).

**219)** More than one reference is given as the source of this value.

**220)** Hedgecock et al. (2005) refer to Hedgecock and Pirrone (2004) as the source but this value cannot be found there.

**221)** Clever and Young (1987) recommend the data from Rettich et al. (1981).

**222)** The data from Reichl (1995) were fitted to the 3-parameter equation: $H_s^{cp} = \exp(-133.87728 + 6629.97157/T + 17.62624 \ln(T))$ mol/(m³ Pa) with $T$ in K.

**223)** The data from Scharlin and Battino (1995) were fitted to the 3-parameter equation: $H_s^{cp} = \exp(-206.41168 + 10058.77208/T + 28.34417 \ln(T))$ mol/(m³ Pa) with $T$ in K.



**224)** The data from Shoor et al. (1969) were fitted to the 3-parameter equation: $H_s^{cp} = \exp(-201.05778 + 9920.37989/T + 27.48020 \ln(T))$ mol/(m$^3$ Pa) with $T$ in K.

**225)** The same value was also published in McAuliffe (1963).

**226)** The same value was also published in McAuliffe (1966).

**227)** The data from Morrison and Billett (1952) were fitted to the 3-parameter equation: $H_s^{cp} = \exp(-195.92072 + 9624.37184/T + 26.74976 \ln(T))$ mol/(m$^3$ Pa) with $T$ in K.

**228)** The data from Winkler (1901) were fitted to the 3-parameter equation: $H_s^{cp} = \exp(-203.15902 + 9951.75251/T + 27.82679 \ln(T))$ mol/(m$^3$ Pa) with $T$ in K.

**229)** Yao et al. (2002) compared two QSPR methods and found that radial basis function networks (RBFNs) are better than multiple linear regression. In their paper, they provide neither a definition nor the unit of their Henry's law constants. Comparing the values with those that they cite from Yaws (1999), it is assumed that they use the variant $H_v^{px}$ and the unit atm.

**230)** English and Carroll (2001) provide several calculations. Here, the preferred value with explicit inclusion of hydrogen bonding parameters from a neural network is shown.

**231)** Value from the training dataset.

**232)** Calculated with a principal component analysis (PCA), see Suzuki et al. (1992) for details.

**233)** The data from Dean and Lange (1999) were fitted to the 3-parameter equation: $H_s^{cp} = \exp(-185.72813 + 9197.97387/T + 25.21142 \ln(T))$ mol/(m$^3$ Pa) with $T$ in K.

**234)** The data from Reichl (1995) were fitted to the 3-parameter equation: $H_s^{cp} = \exp(-109.51433 + 6313.03876/T + 13.60483 \ln(T))$ mol/(m$^3$ Pa) with $T$ in K.

**235)** The data from Morrison and Billett (1952) were fitted to the 3-parameter equation: $H_s^{cp} = \exp(-215.51394 + 10861.98666/T + 29.50128 \ln(T))$ mol/(m$^3$ Pa) with $T$ in K.

**236)** The data from Winkler (1901) were fitted to the 3-parameter equation: $H_s^{cp} = \exp(-277.60377 + 13887.90452/T + 38.63046 \ln(T))$ mol/(m$^3$ Pa) with $T$ in K.

**237)** Value given here as quoted by Gharagheizi et al. (2010).

**238)** Calculated using linear free energy relationships (LFER).

**239)** Calculated using SPARC Performs Automated Reasoning in Chemistry (SPARC).

**240)** Calculated using COSMOtherm.

**241)** Temperature is not specified.

**242)** Value from the training dataset.

**243)** Calculated using the GROMHE model.

**244)** Calculated using the SPARC approach.

**245)** Calculated using the HENRYWIN method.

**246)** Calculated using a combination of a group contribution method and neural networks.

**247)** Modarresi et al. (2005) use different descriptors for their calculations. They conclude that COSA and ANN are the best QSPR models, but COSA is not ideal for hydrocarbons with low solubility. Only results obtained with ANN are shown here.

**248)** Yaffe et al. (2003) present QSPR results calculated with the fuzzy ARTMAP (FAM) and with the back-propagation (BK-Pr) method. They conclude that FAM is better. Only the FAM results are shown here.

**249)** Value from the training set.

**250)** The data from Dean and Lange (1999) were fitted to the 3-parameter equation: $H_s^{cp} = \exp(-249.13770 + 12672.58357/T + 34.34947 \ln(T))$ mol/(m$^3$ Pa) with $T$ in K.

**251)** The data from Reichl (1995) were fitted to the 3-parameter equation: $H_s^{cp} = \exp(-275.67877 + 14048.75446/T + 38.16041 \ln(T))$ mol/(m$^3$ Pa) with $T$ in K.

**252)** The data from Morrison and Billett (1952) were fitted to the 3-parameter equation: $H_s^{cp} = \exp(-257.69118 + 13189.22089/T + 35.51019 \ln(T))$ mol/(m$^3$ Pa) with $T$ in K.

**253)** The H298 and A,B,C data listed in Table 5-4 of Burkholder et al. (2019) are inconsistent, with 14 % difference.

**254)** The H298 and A,B,C data listed in Table 5-4 of Burkholder et al. (2015) are inconsistent, with 14 % difference.

**255)** The H298 and A,B,C data listed in Table 5.4 of Sander et al. (2011) are inconsistent, with 14 % difference.

**256)** The H298 and A,B,C data listed in Table 5.4 of Sander et al. (2006) are inconsistent, with 14 % difference.

**257)** The data from Morrison and Billett (1952) were fitted to the 3-parameter equation: $H_s^{cp} = \exp(-257.40529 + 13425.82235/T + 35.27658 \ln(T))$ mol/(m$^3$ Pa) with $T$ in K.

**258)** Value given here as quoted by Dupeux et al. (2022).



**259)** Calculated using the COSMO-RS method.

**260)** Value from the validation dataset.

**261)** The H298 and A,B,C data listed in Table 5-4 of Burkholder et al. (2019) are inconsistent, with 6 % difference.

**262)** The H298 and A,B,C data listed in Table 5-4 of Burkholder et al. (2015) are inconsistent, with 6 % difference.

**263)** The H298 and A,B,C data listed in Table 5.4 of Sander et al. (2011) are inconsistent, with 6 % difference.

**264)** The H298 and A,B,C data listed in Table 5.4 of Sander et al. (2006) are inconsistent, with 6 % difference.

**265)** Fogg and Sangster (2003) cite an incorrect fitting function from Hayduk (1986).

**266)** The fitting function and the data in the table on page 34 of Hayduk (1986) are inconsistent by a factor of about three. A comparison with the original measurements by Wetlaufer et al. (1964) shows that the data in the table are correct. Refitting the data suggests that the third fitting parameter should be 52.4651, not 53.4651.

**267)** Value from the test set.

**268)** Values at 298 K in Table C2 and C5 of Brockbank (2013) are inconsistent, with 15 % difference.

**269)** The data from Jou and Mather (2000) were fitted to the 3-parameter equation: $H_s^{cp} = \exp(-400.38105 + 20169.61328/T + 56.35286 \ln(T))$ mol/(m³ Pa) with $T$ in K.

**270)** Jou and Mather (2000) also contains high-temperature data. However, only data up to 373.2 K were used here to calculate the temperature dependence.

**271)** Value from the validation dataset.

**272)** Value from the test set.

**273)** The data from Shoor et al. (1969) were fitted to the 3-parameter equation: $H_s^{cp} = \exp(-311.59148 + 15699.27148/T + 43.32183 \ln(T))$ mol/(m³ Pa) with $T$ in K.

**274)** Value from the test dataset.

**275)** Values at 298 K in Table C2 and C5 of Brockbank (2013) are inconsistent, with 6 % difference.

**276)** Apparently, the values in Table 2 of Park et al. (1997) show $\log_{10}(K_{aw})$ and not $K_{aw}$ as their figure caption states.

**277)** Extrapolated from data measured between 40 °C and 80 °C.

**278)** Data are taken from the report by Howe et al. (1987).

**279)** Value from the training set.

**280)** In their Table 8, Staudinger and Roberts (1996) incorrectly cite a value given by Ashworth et al. (1988).

**281)** The same data were also published in Hansen et al. (1995).

**282)** Hansen et al. (1993) found that the solubility of 2-methylhexane increases with temperature.

**283)** Values at 298 K in Table C2 and C5 of Brockbank (2013) are inconsistent, with 5 % difference.

**284)** The data from Dohányosová et al. (2004) were fitted to the 3-parameter equation: $H_s^{cp} = \exp(-670.94997 + 33188.34075/T + 95.95541 \ln(T))$ mol/(m³ Pa) with $T$ in K.

**285)** Values at 298 K in Table C2 and C5 of Brockbank (2013) are inconsistent, with 21 % difference.

**286)** The data from Dohányosová et al. (2004) were fitted to the 3-parameter equation: $H_s^{cp} = \exp(-792.29258 + 38089.35992/T + 114.36667 \ln(T))$ mol/(m³ Pa) with $T$ in K.

**287)** Data taken from the supplement.

**288)** Calculated using the EPI Suite (v4.0) method.

**289)** Calculated using the SPARC (v4.2) method.

**290)** Calculated using the COSMOtherm (v2.1) method.

**291)** Calculated using the ABSOLV (ADMEBoxes v4.1) method.

**292)** Mackay et al. (2006a) list a vapor pressure $p$, a solubility $c$, and a Henry's law constant calculated as $p/c$. However, the data are internally inconsistent and deviate by more than 10 %.

**293)** Values at 298 K in Table C2 and C5 of Brockbank (2013) are inconsistent, with 23 % difference.

**294)** Value at $T = 294$ K.

**295)** Values at 298 K in Table C2 and C5 of Brockbank (2013) are inconsistent, with 13 % difference.

**296)** The data listed in Tabs. 2 and 3 of Dewulf et al. (1999) are inconsistent, with 5 % difference.

**297)** Value at $T = 301$ K.

**298)** Value given here as quoted by Staudinger and Roberts (1996).

**299)** Value from the test set for true external validation.

**300)** The data from Dohányosová et al. (2004) were fitted to the 3-parameter equation: $H_s^{cp} = \exp(-365.40645 + 19821.40051/T + 50.78223 \ln(T))$ mol/(m³ Pa) with $T$ in K.

**301)** The data from Dohányosová et al. (2004) were fitted to the 3-parameter equation: $H_s^{cp} = \exp(-383.72514 + 20514.87228/T + 53.42859 \ln(T))$ mol/(m³ Pa) with $T$ in K.



**302)** The data from Dohányosová et al. (2004) were fitted to the 3-parameter equation: $H_s^{cp} = \exp(-369.42853 + 19642.40603/T + 51.34116 \ln(T))$ mol/(m$^3$ Pa) with $T$ in K.

**303)** Haynes (2014) refer to Mackay and Shiu (1981) but that article lists this value for 1,4-dimethylcyclohexane, not for 1,2-dimethylcyclohexane.

**304)** Values at 298 K in Table C2 and C5 of Brockbank (2013) are inconsistent, with 6 % difference.

**305)** The data from Dohányosová et al. (2004) were fitted to the 3-parameter equation: $H_s^{cp} = \exp(-346.32561 + 18710.63122/T + 47.87398 \ln(T))$ mol/(m$^3$ Pa) with $T$ in K.

**306)** The data from Maaßen (1995) were fitted to the 3-parameter equation: $H_s^{cp} = \exp(-187.57836 + 9639.75245/T + 25.50544 \ln(T))$ mol/(m$^3$ Pa) with $T$ in K.

**307)** The data from Reichl (1995) were fitted to the 3-parameter equation: $H_s^{cp} = \exp(-166.44394 + 8613.39266/T + 22.39721 \ln(T))$ mol/(m$^3$ Pa) with $T$ in K.

**308)** The data from Morrison and Billett (1952) were fitted to the 3-parameter equation: $H_s^{cp} = \exp(-175.14997 + 9028.26949/T + 23.67675 \ln(T))$ mol/(m$^3$ Pa) with $T$ in K.

**309)** The data from Dean and Lange (1999) were fitted to the 3-parameter equation: $H_s^{cp} = \exp(-221.00286 + 11107.47493/T + 30.50401 \ln(T))$ mol/(m$^3$ Pa) with $T$ in K.

**310)** The data from Maaßen (1995) were fitted to the 3-parameter equation: $H_s^{cp} = \exp(-168.51157 + 9378.22622/T + 22.33127 \ln(T))$ mol/(m$^3$ Pa) with $T$ in K.

**311)** The data from Reichl (1995) were fitted to the 3-parameter equation: $H_s^{cp} = \exp(-241.54655 + 12718.75981/T + 33.18333 \ln(T))$ mol/(m$^3$ Pa) with $T$ in K.

**312)** The data from Serra and Palavra (2003) were fitted to the 3-parameter equation: $H_s^{cp} = \exp(-261.78355 + 13728.91505/T + 36.10688 \ln(T))$ mol/(m$^3$ Pa) with $T$ in K.

**313)** According to Donahue and Prinn (1993), the value is incorrect.

**314)** Wang et al. (2017) provide separate data for *cis* and *trans*. However, since both isomers are identified by the same SMILES string in their study, it is unclear how the stereochemistry has been taken into account.

**315)** Values for the Henry's law constants shown in Fig. 3 of Martins et al. (2017) were obtained from S. Pinho (pers. comm., 2022).

**316)** Ebert et al. (2023) present "curated experimental" Henry's law constants from the literature but do not provide any references. It is only mentioned that the value is from multiple sources and obtained from experimental vapor pressure and water solubility.

**317)** The data from Dohányosová et al. (2004) were fitted to the 3-parameter equation: $H_s^{cp} = \exp(-169.70973 + 10843.51763/T + 21.91320 \ln(T))$ mol/(m$^3$ Pa) with $T$ in K.

**318)** Ebert et al. (2023) present "curated experimental" Henry's law constants from the literature but do not provide any references. It is only mentioned that the value is from a single database or data collection and obtained from experimental vapor pressure and water solubility.

**319)** Approximate value extracted from Fig. 1 of Maillard and Rosenthal (1952).

**320)** The same article was also published in Monatshefte für Chemie 23, 489–501 (1902).

**321)** Value given here as quoted by Fogg et al. (2001).

**322)** The data from Dean and Lange (1999) were fitted to the 3-parameter equation: $H_s^{cp} = \exp(-143.25283 + 7542.89338/T + 19.33269 \ln(T))$ mol/(m$^3$ Pa) with $T$ in K.

**323)** Regression and individual data points of Simpson and Lovell (1962) are inconsistent, with 5 % difference.

**324)** Using the theoretical initial concentration (H0), see Zhang et al. (2013) for details.

**325)** Average of all duplicates (H1), see Zhang et al. (2013) for details.

**326)** Sieg et al. (2009) also provide data for supercooled water. Here, only data above 0 °C were used to calculate the temperature dependence.

**327)** Extrapolated from data above 298 K.

**328)** It was found that $H_s$ changes with the concentration of the solution.

**329)** The data from Görgényi et al. (2002) were fitted to the 3-parameter equation: $H_s^{cp} = \exp(-346.88030 + 18421.52810/T + 48.91393 \ln(T))$ mol/(m$^3$ Pa) with $T$ in K.

**330)** Value obtained by applying a modified batch air-stripping method, otherwise called the vapor entry loop (VEL) method, see Kochetkov et al. (2001) for details.

**331)** Value obtained by applying the static head space (HS) method, see Kochetkov et al. (2001) for details.

**332)** The data from Khalfaoui and Newsham (1994b) were fitted to the 3-parameter equation: $H_s^{cp} = \exp(-129.36095 + 8999.48627/T + 16.29087 \ln(T))$ mol/(m$^3$ Pa) with $T$ in K.



**333)** The data from Robbins et al. (1993) were fitted to the 3-parameter equation: $H_s^{cp} = \exp(189.41389 - 5855.10843/T - 30.90289\ln(T))$ mol/(m³ Pa) with $T$ in K.

**334)** Value at $T = 302$ K.

**335)** The data from Cooling et al. (1992) were fitted to the 3-parameter equation: $H_s^{cp} = \exp(-231.38331 + 13640.47358/T + 31.46504\ln(T))$ mol/(m³ Pa) with $T$ in K.

**336)** Calculated using $G_h$ and $H_h$ from Table 2 in Andon et al. (1954). Note that the thermodynamic functions in that table are not based on their $\alpha$ in Table 1. Instead, the expression $\exp(-G_h/(RT))$ yields the Henry's law constant $H_s^{xp}$ in the unit 1/atm.

**337)** Values at 298 K in Table C2 and C5 of Brockbank (2013) are inconsistent, with 11 % difference.

**338)** Values for salt solutions are also available from this reference.

**339)** The data from Görgényi et al. (2002) were fitted to the 3-parameter equation: $H_s^{cp} = \exp(-468.28203 + 24099.39947/T + 66.85565\ln(T))$ mol/(m³ Pa) with $T$ in K.

**340)** Value obtained by applying the EPICS method, see Ayuttaya et al. (2001) for details.

**341)** Value obtained by applying the static cell (linear form) method, see Ayuttaya et al. (2001) for details.

**342)** Value obtained by applying the direct phase concentration ratio method, see Ayuttaya et al. (2001) for details.

**343)** Value obtained by applying the static cell (nonlinear form) method, see Ayuttaya et al. (2001) for details.

**344)** The data from Robbins et al. (1993) were fitted to the 3-parameter equation: $H_s^{cp} = \exp(-573.76928 + 28956.65188/T + 82.51911\ln(T))$ mol/(m³ Pa) with $T$ in K.

**345)** The temperature dependence is recalculated using the data in Table 4 of Lamarche and Droste (1989) and not taken from their Table 5.

**346)** Apparently, the vapor pressure of toluene was used to calculate its Henry's law constant. However, no source is provided.

**347)** Value given here as quoted by Dewulf et al. (1995).

**348)** The data from Robbins et al. (1993) were fitted to the 3-parameter equation: $H_s^{cp} = \exp(-1350.74178 + 64760.28328/T + 197.85937\ln(T))$ mol/(m³ Pa) with $T$ in K.

**349)** The data from Schwardt et al. (2021) were fitted to the 3-parameter equation: $H_s^{cp} = \exp(100.47045 - 2603.76722/T - 17.31043\ln(T))$ mol/(m³ Pa) with $T$ in K.

**350)** Value given here as quoted by HSDB (2015).

**351)** The regression parameters for ethylbenzene in Tab. 1 of Schwardt et al. (2021) are wrong. Corrected values from Schwardt et al. (2022) are used here.

**352)** The data from Schwardt et al. (2021) were fitted to the 3-parameter equation: $H_s^{cp} = \exp(-176.88587 + 11290.74921/T + 23.22869\ln(T))$ mol/(m³ Pa) with $T$ in K.

**353)** Different types of Henry's law constants of Ryu and Park (1999) are inconsistent, with 14 % difference.

**354)** The data from Robbins et al. (1993) were fitted to the 3-parameter equation: $H_s^{cp} = \exp(-371.46947 + 20514.07888/T + 51.95086\ln(T))$ mol/(m³ Pa) with $T$ in K.

**355)** The value listed as $A$ for diethylbenzene is probably not $A$ but the Henry's law volatility constant $H_v^{px}$ at 298 K.

**356)** Yaffe et al. (2003) list this species twice in their table, with different values. As it is unclear which is correct, the data are not reproduced here.

**357)** Erratum for page 365 of Fogg and Sangster (2003): Data from Kondoh and Nakajima (1997) are cited incorrectly, giving the same values at 308.2 K and 318.2 K, respectively.

**358)** Value from the external prediction set.

**359)** Values at 298 K in Table C2 and C5 of Brockbank (2013) are inconsistent, with 8 % difference.

**360)** Because of discrepancies between the values shown in Tables 4 and 5 of Shiu and Ma (2000), the data are not used here.

**361)** Effective Henry's law constants at several pH values are provided by van Ruth and Villeneuve (2002). Here, only the value at pH 3 is shown.

**362)** The values of Dewulf et al. (1999) are not used here because, according to them, the calculated regression does not match the theoretical expectation for this species.

**363)** Calculated using the COSMO-RS method.

**364)** Value given here as quoted by Haynes (2014).

**365)** Ebert et al. (2023) present "curated experimental" Henry's law constants from the literature but do not provide any references. It is only mentioned that the value is from a single database or data collection and measured directly.

**366)** Literature-derived value.

**367)** Final adjusted value.

**368)** Value given here as quoted by Petrasek et al. (1983).

**369)** Calculated using COSMOtherm.

**370)** Calculated using the COSMO-RS method.




**371)** Ebert et al. (2023) present "curated experimental" Henry's law constants from the literature but do not provide any references. It is only mentioned that the value is from a single original paper and obtained from experimental vapor pressure and water solubility.

**372)** Value at $T = 299$ K.

**373)** Value at $T = 283$ K.

**374)** Cargill (1990) recommends the data from Rettich et al. (1982).

**375)** The data from Rettich et al. (1982) were fitted to the 3-parameter equation: $H_s^{cp} = \exp(-188.21737 + 8974.05844/T + 25.72558 \ln(T))$ mol/(m$^3$ Pa) with $T$ in K.

**376)** The data from Douglas (1967) were fitted to the 3-parameter equation: $H_s^{cp} = \exp(-180.92848 + 8514.05914/T + 24.68060 \ln(T))$ mol/(m$^3$ Pa) with $T$ in K.

**377)** Solubility in sea water at 20.99 % chlorinity.

**378)** The data from Winkler (1901) were fitted to the 3-parameter equation: $H_s^{cp} = \exp(-163.07031 + 7890.85881/T + 21.94517 \ln(T))$ mol/(m$^3$ Pa) with $T$ in K.

**379)** The data from Dean and Lange (1999) were fitted to the 3-parameter equation: $H_s^{cp} = \exp(-161.93492 + 7852.78262/T + 21.76812 \ln(T))$ mol/(m$^3$ Pa) with $T$ in K.

**380)** The data from Zheng et al. (1997) were fitted to the 3-parameter equation: $H_s^{cp} = \exp(-144.44443 + 8071.06186/T + 19.20040 \ln(T))$ mol/(m$^3$ Pa) with $T$ in K.

**381)** The data from Murray and Riley (1971) were fitted to the 3-parameter equation: $H_s^{cp} = \exp(-167.86941 + 9146.24434/T + 22.67331 \ln(T))$ mol/(m$^3$ Pa) with $T$ in K.

**382)** The data from Morrison and Billett (1952) were fitted to the 3-parameter equation: $H_s^{cp} = \exp(-126.83009 + 7302.88179/T + 16.55553 \ln(T))$ mol/(m$^3$ Pa) with $T$ in K.

**383)** The data from Bohr (1899) were fitted to the 3-parameter equation: $H_s^{cp} = \exp(-140.70007 + 7951.73013/T + 18.60961 \ln(T))$ mol/(m$^3$ Pa) with $T$ in K.

**384)** As mentioned by Fogg and Sangster (2003), the fitting equation by Scharlin (1996) is erroneous. It appears that a correction factor of about $10^6$ is necessary for consistency with their own data in Tab. 1.

**385)** The data from Dean and Lange (1999) were fitted to the 3-parameter equation: $H_s^{cp} = \exp(-138.54120 + 7859.16351/T + 18.28486 \ln(T))$ mol/(m$^3$ Pa) with $T$ in K.

**386)** Keßel et al. (2017) provide data at several pH values. Here, only the value at pH 2 is shown because hydrolyses occurs in more alkaline solutions.

**387)** The H298 and A,B,C data listed in Table 5.4 of Sander et al. (2011) are inconsistent, with 92 % difference.

**388)** This paper supersedes earlier work with more concentrated solutions (Butler et al., 1933).

**389)** Value given here as quoted by Gaffney et al. (1987).

**390)** Value given here as quoted by Hine and Weimar (1965).

**391)** The H298 and A,B,C data listed in Table 5-4 of Burkholder et al. (2019) are inconsistent, with 10 % difference.

**392)** The H298 and A,B,C data listed in Table 5-4 of Burkholder et al. (2015) are inconsistent, with 10 % difference.

**393)** Values at 298 K in Table C2 and C5 of Brockbank (2013) are inconsistent, with 8 % difference.

**394)** The H298 and A,B,C data listed in Table 5.4 of Sander et al. (2011) are inconsistent, with 10 % difference.

**395)** Extrapolated from data above 298 K.

**396)** Koga (1995) found that *tert*-butanol does not obey Henry's law at $c > 3.8$ mM.

**397)** Incorrect data are given by Burkholder et al. (2019) for 2-methyl-2-propanol. The correct parameter for the temperature dependence is: C = 37.98 (R. E. Huie, pers. comm. 2021).

**398)** Incorrect data are given by Burkholder et al. (2015) for 2-methyl-2-propanol. The correct parameter for the temperature dependence is: C = 37.98 (R. E. Huie, pers. comm. 2021).

**399)** Calculated for an aqueous solution containing 60 % ethanol by volume as the solvent.

**400)** Values at 298 K in Table C2 and C5 of Brockbank (2013) are inconsistent, with 6 % difference.

**401)** Value obtained by Saxena and Hildemann (1996) using the group contribution method.

**402)** Value at $T = 300$ K.

**403)** The error given by Suzuki et al. (1992) is not the difference between the observed and the calculated value, as it should be. It is unclear which of the numbers is wrong.

**404)** The species is probably 2,3-dimethyl-2-butanol and not 2,3-dimethylbutanol as listed in Hine and Mookerjee (1975).

**405)** Rumble (2021) refers to Moore et al. (1995) as the source but this value cannot be found there.



**406)** It is assumed here that entry number 72 in Table 1 of Yaws et al. (1997) refers to 2-methyl-1-heptanol, not 2-methyl-2-heptanol.

**407)** KWAC and KAWp from Table 2 of Lei et al. (2007) are inconsistent, with 10 % difference.

**408)** Values at 298 K in Table C2 and C5 of Brockbank (2013) are inconsistent, with 8 % difference.

**409)** Different types of Henry's law constants of Yaws and Yang (1992) are inconsistent, with 16 % difference.

**410)** Different types of Henry's law constants of Yaws and Yang (1992) are inconsistent, with 10 % difference.

**411)** Value at $T$ = 307 K.

**412)** Value given here as quoted by Mackay et al. (1995).

**413)** Calculated using SPARC Performs Automated Reasoning in Chemistry (SPARC). It is assumed here that the value refers to $T$ = 298.15 K.

**414)** Value given here as quoted by Hine and Mookerjee (1975).

**415)** Value at $T$ = 373 K.

**416)** Value at $T$ = 281 K.

**417)** It is assumed here that the thermodynamic data refers to the units $[\mathrm{mol\,dm^{-3}}]$ and $[\mathrm{atm}]$ as standard states.

**418)** Value given here as quoted by Shiu et al. (1994).

**419)** HSDB (2015) refer to Abraham et al. (1994b) as the source but this value cannot be found there. Maybe the value is taken from Abraham et al. (1990).

**420)** Mackay et al. (2006c) list a vapor pressure $p$, a solubility $c$, and a Henry's law constant calculated as $p/c$. However, the data are internally inconsistent and deviate by more than 10 %.

**421)** Betterton (1992) gives Buttery et al. (1969) as the source. However, no data were found in that reference.

**422)** Intermediate of estimates employing the bond method from the EPI HENRYWIN software.

**423)** Although Mansfield (2020) writes that his "Table 6 summarizes numerical calculations for formaldehyde and acetaldehyde assuming the values given in Tables 4 and 5", different values for the Henry's law constants are shown in these tables.

**424)** Saxena and Hildemann (1996) say that this value is unreliable.

**425)** Calculated using the free energy perturbation (FEP) method.

**426)** Calculated using the thermodynamic integration (TI) method.

**427)** Calculated using the Bennett acceptance ratio (BAR) method.

**428)** Saxena and Hildemann (1996) give a range of $9.9 \times 10^2$ mol/(m$^3$ Pa) $< H_s^{cp} < 5.9 \times 10^4$ mol/(m$^3$ Pa).

**429)** Saxena and Hildemann (1996) give a range of $5.9 \times 10^6$ mol/(m$^3$ Pa) $< H_s^{cp} < 3.9 \times 10^9$ mol/(m$^3$ Pa).

**430)** The formula of 1,2-butanediol is incorrectly given as "HOCH(OH)C$_2$H$_5$" by Burkholder et al. (2019).

**431)** The formula of 1,2-butanediol is incorrectly given as "HOCH(OH)C$_2$H$_5$" by Burkholder et al. (2015).

**432)** Saxena and Hildemann (1996) give a range of $9.9 \times 10^2$ mol/(m$^3$ Pa) $< H_s^{cp} < 4.9 \times 10^4$ mol/(m$^3$ Pa).

**433)** Saxena and Hildemann (1996) give a range of $3.9 \times 10^2$ mol/(m$^3$ Pa) $< H_s^{cp} < 3.9 \times 10^4$ mol/(m$^3$ Pa).

**434)** Calculated based on atmospheric measurements.

**435)** Calculated using EPI.

**436)** Calculated using SPARC.

**437)** Henry's law constants calculated using the GROMHE model. Temperature-dependences calculated with the method of Kühne et al. (2005).

**438)** Isaacman-VanWertz et al. (2016) refer to Raventos-Duran et al. (2010) as the source but the quoted value cannot be found there.

**439)** Calculated using GROMHE.

**440)** Isaacman-VanWertz et al. (2016) refer to a paper by Hilal et al. as the source but the quoted value cannot be found there.

**441)** Calculated using SPARC.

**442)** Calculated using the bond contribution of HENRYWIN.

**443)** Compernolle and Müller (2014b) recommend $H_s^{cp}$ for 1,7-heptanediol in the range of $4.5 \times 10^4$ mol/(m$^3$ Pa) $< H_s^{cp} < 8.3 \times 10^4$ mol/(m$^3$ Pa).

**444)** Compernolle and Müller (2014b) recommend $H_s^{cp}$ for 1,9-nonanediol in the range of $2.4 \times 10^4$ mol/(m$^3$ Pa) $< H_s^{cp} < 3.9 \times 10^4$ mol/(m$^3$ Pa).

**445)** Compernolle and Müller (2014b) recommend $H_s^{cp}$ for 1,10-decanediol in the range of $2.5 \times 10^4$ mol/(m$^3$ Pa) $< H_s^{cp} < 3.0 \times 10^4$ mol/(m$^3$ Pa).

**446)** Value given here as quoted by Hilal et al. (2008).

**447)** Calculated using the EPI Suite method (https://www.epa.gov/tsca-screening-tools/epi-suitetm-estimation-program-interface).

**448)** Value for the temperature range from 261 K to 281 K.

**449)** Value at $T$ = 278 K.



**450)** Leriche et al. (2000) assume $H_s(\mathrm{ROO}) = H_s(\mathrm{ROOH}) \times H_s(\mathrm{HO_2})/H_s(\mathrm{H_2O_2})$.

**451)** Lelieveld and Crutzen (1991) assume $H_s(\mathrm{CH_3OO}) = H_s(\mathrm{HO_2})$.

**452)** Jacob (1986) assumes $H_s(\mathrm{CH_3OO}) = H_s(\mathrm{CH_3OOH}) \times H_s(\mathrm{HO_2})/H_s(\mathrm{H_2O_2})$.

**453)** Calculated using EVAPORATION and AIOMFAC.

**454)** Calculated using the GROMHE model.

**455)** Effective value that takes into account the hydration of HCHO:

$$H_s = ([\mathrm{HCHO}] + [\mathrm{CH_2(OH)_2}])/p(\mathrm{HCHO})$$

**456)** Data from Table 1 by Zhou and Mopper (1990) were used to redo the regression analysis. The data for acetone in their Table 2 are incorrect.

**457)** Dong and Dasgupta (1986) found that the Henry's law constant for HCHO is not a true constant but increases with increasing concentration. Note that their expression does not converge asymptotically to a constant value at infinite dilution.

**458)** Ledbury and Blair (1925) (and also Blair and Ledbury (1925)) measured the solubility of HCHO at very high concentrations around 5 to 15 M. Their value of $H_s$ increases with HCHO concentration. Lelieveld and Crutzen (1991), Hough (1991), and Pandis and Seinfeld (1989) all use these solubility data but do not specify how they extrapolated to lower concentrations. Since the concentration range is far from typical values in atmospheric chemistry, the value is not reproduced here.

**459)** Value given here as quoted by Möller and Mauersberger (1992).

**460)** Effective value that takes into account the hydration of the aldehyde:

$$H_s = ([\mathrm{RCHO}] + [\mathrm{RCH(OH)_2}])/p(\mathrm{RCHO})$$

**461)** The data from Wieland et al. (2015) were fitted to the 3-parameter equation: $H_s^{cp} = \exp(25.01220 + 3596.11696/T - 6.81730\ln(T))$ mol/(m³ Pa) with $T$ in K.

**462)** Value given here as quoted by Bone et al. (1983).

**463)** Value suitable for the conditions of a case study in Mexico City.

**464)** Volkamer et al. (2009) found average effective Henry's law constants for CHOCHO in the range $1.6{\times}10^6$ mol/(m³ Pa) $< H_s^{cp} < 5.9{\times}10^6$ mol/(m³ Pa) for solutions containing ammonium sulfate and/or fulvic acid. A salting-in effect by fulvic acid was observed even in the absence of sulfate.

**465)** Solubility in sulfate aerosol.

**466)** Woo and McNeill (2015) say that the Henry's law constant was updated based on advances in the literature since McNeill et al. (2012) but do not provide further details.

**467)** Value at $T$ = 372 K.

**468)** The data from Wieland et al. (2015) were fitted to the 3-parameter equation: $H_s^{cp} = \exp(-176.35942 + 12895.73116/T + 22.70566\ln(T))$ mol/(m³ Pa) with $T$ in K.

**469)** The formula of propenal is incorrectly given as "CH₂CHO" by Burkholder et al. (2019).

**470)** The temperature dependence parameter C for 2-butenal is missing in Burkholder et al. (2019). The correct value is: C = 24.42 (R. E. Huie, pers. comm. 2021).

**471)** The data from Buttery et al. (1971) for trans-2-octenal are incorrectly cited by Betterton (1992).

**472)** Values at 298 K in Table C2 and C5 of Brockbank (2013) are inconsistent, with 6 % difference.

**473)** Calculated under the assumption that $\Delta G$ and $\Delta H$ are based on [mol/l] and [atm] as the standard states.

**474)** Calculated using the experimental value adjusted (EVA) method, see McFall et al. (2020) for details.

**475)** Value at $T$ = 359 K.

**476)** Values at 298 K in Table C2 and C5 of Brockbank (2013) are inconsistent, with 5 % difference.

**477)** Calculated from the slope of $y_{\mathrm{ac}}P$ vs $x_{\mathrm{ac}}$, using data from Tab. VIII in Lichtenbelt and Schram (1985).

**478)** Value at $T$ = 313 K.

**479)** Values at 298 K in Table C2 and C5 of Brockbank (2013) are inconsistent, with 6 % difference.

**480)** Values at 298 K in Table C2 and C5 of Brockbank (2013) are inconsistent, with 7 % difference.

**481)** Values at 298 K in Table C2 and C5 of Brockbank (2013) are inconsistent, with 6 % difference.

**482)** Table S2 in the supplement of Wu et al. (2022a) contains incorrect data for 3-octanone. Here, the corrected data (S. Wu, pers. comm. 2022) were used: 2.88E-2 and 1.52E-2 at 25 °C and 35 °C, respectively.

**483)** Values at 298 K in Table C2 and C5 of Brockbank (2013) are inconsistent, with 7 % difference.

**484)** The value listed as $A$ for 2,6,8-trimethyl-4-nonanone is probably not $A$ but the Henry's law volatility constant $H_v^{px}$ at 298 K.

**485)** The data from Wieland et al. (2015) were fitted to the 3-parameter equation: $H_s^{cp} = \exp(116.85779 - 1341.05519/T - 19.91967\ln(T))$ mol/(m³ Pa) with $T$ in K.



**486)** The data from Wieland et al. (2015) were fitted to the 3-parameter equation: $H_s^{cp} = \exp(-74.84087 + 9452.88617/T + 7.41865 \ln(T))$ mol/(m$^3$ Pa) with $T$ in K.

**487)** The value given here was measured at a liquid phase mixing ratio of 1 µmol/mol. Servant et al. (1991) found that the Henry's law constant changes at higher concentrations.

**488)** Abraham (1984) smoothed the values from a plot of enthalpy against carbon number.

**489)** The value of $H_s^{\ominus}$ was taken from Keene and Galloway (1986).

**490)** Calculated using thermodynamic data from Latimer (1952).

**491)** Value at pH = 4.

**492)** Calculated using HENRYWIN 3.2 (bond contribution method).

**493)** At pH 7.

**494)** At pH 10.8.

**495)** Value at $T$ = 289 K.

**496)** Value at $T$ = 338 K.

**497)** Pecsar and Martin (1966) is quoted as the source. However, only activity coefficients and no vapor pressures are listed there.

**498)** The H298 and A,B,C data listed in Table 5-4 of Burkholder et al. (2019) are inconsistent, with 6 % difference.

**499)** The formula of methyl ethanoate is incorrectly given as "CH$_3$C(O)CH$_3$" by Burkholder et al. (2015).

**500)** The H298 and A,B data listed in Table 5-4 of Burkholder et al. (2015) are inconsistent, with 74 % difference.

**501)** The same data were also published in Kieckbusch and King (1979a).

**502)** The H298 and A,B,C data listed in Table 5-4 of Burkholder et al. (2019) are inconsistent, with 7 % difference.

**503)** The formula of propyl ethanoate is incorrectly given as "CH$_3$C(O)C$_3$H$_8$" by Burkholder et al. (2019).

**504)** Katritzky et al. (1998) list this species twice in their table, with different values. As it is unclear which of them is correct, the data are not reproduced here.

**505)** Values at 298 K in Table C2 and C5 of Brockbank (2013) are inconsistent, with 5 % difference.

**506)** Values at 298 K in Table C2 and C5 of Brockbank (2013) are inconsistent, with 6 % difference.

**507)** The value listed as $A$ for n-heptyl acetate is probably not $A$ but the Henry's law volatility constant $H_v^{px}$ at 298 K.

**508)** The value listed as $A$ for n-octyl acetate is probably not $A$ but the Henry's law volatility constant $H_v^{px}$ at 298 K.

**509)** Betterton (1992) gives Kieckbusch and King (1979b) as the source. However, no data were found in that reference.

**510)** Values at 298 K in Table C2 and C5 of Brockbank (2013) are inconsistent, with 6 % difference.

**511)** Values at 298 K in Table C2 and C5 of Brockbank (2013) are inconsistent, with 5 % difference.

**512)** Values at 298 K in Table C2 and C5 of Brockbank (2013) are inconsistent, with 6 % difference.

**513)** The data from Wieland et al. (2015) were fitted to the 3-parameter equation: $H_s^{cp} = \exp(34.46832 + 3269.29552/T - 8.76905 \ln(T))$ mol/(m$^3$ Pa) with $T$ in K.

**514)** Burkholder et al. (2019) refer to Dohnal et al. (2010) but the quoted value cannot be found there.

**515)** Burkholder et al. (2015) refer to Dohnal et al. (2010) but the quoted value cannot be found there.

**516)** Dipropyl phthalate is listed twice with different values.

**517)** Values at 298 K in Table C2 and C5 of Brockbank (2013) are inconsistent, with 9 % difference.

**518)** Values at 298 K in Table C2 and C5 of Brockbank (2013) are inconsistent, with 17 % difference.

**519)** Hwang et al. (2010) present regression parameters in their Tab. 6 and values extrapolated to 298.15 K in their Tab. 5. However, I was not able to reproduce their calculation. The data shown here are from my own regression of the measured data between 318.15 K and 333.15 K.

**520)** The data from Haimi et al. (2006) were fitted to the 3-parameter equation: $H_s^{cp} = \exp(752.39274 - 29351.83448/T - 115.55407 \ln(T))$ mol/(m$^3$ Pa) with $T$ in K.

**521)** Different types of Henry's law constants of Arp and Schmidt (2004) are inconsistent, with 5 % difference.

**522)** The data from Robbins et al. (1993) were fitted to the 3-parameter equation: $H_s^{cp} = \exp(-4264.16032 + 202439.46180/T + 628.54371 \ln(T))$ mol/(m$^3$ Pa) with $T$ in K.

**523)** The data from Haimi et al. (2006) were fitted to the 3-parameter equation: $H_s^{cp} = \exp(224.10069 - 4205.03828/T - 37.65761 \ln(T))$ mol/(m$^3$ Pa) with $T$ in K.

**524)** The data from Haimi et al. (2006) were fitted to the 3-parameter equation: $H_s^{cp} = \exp(-780.30940 + 40758.59752/T + 112.07468 \ln(T))$ mol/(m$^3$ Pa) with $T$ in K.



**525)** The data from Haimi et al. (2006) were fitted to the 3-parameter equation: $H_s^{cp} = \exp(-565.00561 + 31411.46240/T + 79.73748\ln(T))$ mol/(m$^3$ Pa) with $T$ in K.

**526)** Ebert et al. (2023) present "curated experimental" Henry's law constants from the literature but do not provide any references. It is only mentioned that the value is from a single original paper and obtained from experimental vapor pressure and the infinite-dilution activity coefficient.

**527)** The data from Haimi et al. (2006) were fitted to the 3-parameter equation: $H_s^{cp} = \exp(-1125.52184 + 56732.54277/T + 163.04749\ln(T))$ mol/(m$^3$ Pa) with $T$ in K.

**528)** The data from Haimi et al. (2006) were fitted to the 3-parameter equation: $H_s^{cp} = \exp(-1315.53726 + 64110.36765/T + 191.89554\ln(T))$ mol/(m$^3$ Pa) with $T$ in K.

**529)** The value listed as $A$ for di-$n$-pentyl ether is probably not $A$ but the Henry's law volatility constant $H_v^{px}$ at 298 K.

**530)** The value listed as $A$ for di-$n$-hexyl ether is probably not $A$ but the Henry's law volatility constant $H_v^{px}$ at 298 K.

**531)** Values at 298 K in Table C2 and C5 of Brockbank (2013) are inconsistent, with 5 % difference.

**532)** The data from Wieland et al. (2015) were fitted to the 3-parameter equation: $H_s^{cp} = \exp(-157.10556 + 10203.60762/T + 20.42555\ln(T))$ mol/(m$^3$ Pa) with $T$ in K.

**533)** Betterton (1992) gives Hine and Weimar (1965) as the source. However, no data were found in that reference.

**534)** Betterton (1992) gives Vitenberg et al. (1975) as the source. However, no data were found in that reference.

**535)** Based on gas chromatograph retention indices (GC-RIs).

**536)** Warneck (2005) refers to Saxena and Hildemann (1996) as the source but the quoted value cannot be found there.

**537)** The formula of hydroxyethanoic acid is incorrectly given as "HC(OH)C(O)OH" by Burkholder et al. (2019).

**538)** The formula of hydroxyethanoic acid is incorrectly given as "HC(OH)C(O)OH" by Burkholder et al. (2015).

**539)** Temperature dependencies in Tabs. 1 and 2 of Ashworth et al. (1988) are inconsistent, with 31 % difference.

**540)** Compernolle and Müller (2014a) recommend $H_s^{cp}$ for tartaric acid in the range of $6.9{\times}10^{14}$ mol/(m$^3$ Pa) $< H_s^{cp} < 9.2{\times}10^{15}$ mol/(m$^3$ Pa).

**541)** Chan et al. (2010) give a range of $1.9{\times}10^5$ mol/(m$^3$ Pa) $< H_s^{cp} < 9.5{\times}10^6$ mol/(m$^3$ Pa).

**542)** Calculated using the HENRYWIN program and calibration to 1,3-propanediol.

**543)** The value was chosen for a model study because it gave the best agreement with measurements.

**544)** Center of the range $(2.3\ldots4.0)$ mol/(m$^3$ Pa).

**545)** Calculated based on the method by Hine and Mookerjee (1975).

**546)** Compernolle and Müller (2014a) recommend $H_s^{cp}$ for citric acid in the range of $2.0{\times}10^{14}$ mol/(m$^3$ Pa) $< H_s^{cp} < 5.9{\times}10^{15}$ mol/(m$^3$ Pa).

**547)** The data from Wieland et al. (2015) were fitted to the 3-parameter equation: $H_s^{cp} = \exp(-96.39127 + 11107.87195/T + 10.76466\ln(T))$ mol/(m$^3$ Pa) with $T$ in K.

**548)** In their Fig. 5b, Kish et al. (2013) apply an unspecified factor to the Henry's law constant, and it is not clear if the temperature dependence shown there is correct (Y. Liu, pers. comm. 2014).

**549)** Calculated using the method from Nguyen (2013).

**550)** Calculated from the slope of $y_1 P$ vs $x_1$, using the tabulated VLE data from Kim et al. (2008) between 40 °C and 100 °C. Only dilute solutions with $x_1 \leq 0.1$ were considered.

**551)** Value at $T = 309$ K.

**552)** The data from Christie and Crisp (1967) for dipropylamine are incorrectly cited by Betterton (1992).

**553)** Value at $T = 323$ K.

**554)** Values at 298 K in Table C2 and C5 of Brockbank (2013) are inconsistent, with 6 % difference.

**555)** Value at $T = 308$ K.

**556)** Values at 298 K in Table C2 and C5 of Brockbank (2013) are inconsistent, with 5 % difference.

**557)** Value at $T = 285$ K.

**558)** Mackay et al. (2006d) list a vapor pressure $p$, a solubility $c$, and a Henry's law constant calculated as $p/c$. However, the data are internally inconsistent and deviate by more than 10 %.

**559)** Calculated using $\Delta G_s^{g\rightarrow\text{H}_2\text{O}}$ and $\Delta H_s^{g\rightarrow\text{H}_2\text{O}}$ from Table IV of Arnett and Chawla (1979). Since some of the values in this table are taken directly from Andon et al. (1954), it is assumed that the thermodynamic properties are defined in the same way. Since $\Delta H_s^{g\rightarrow\text{H}_2\text{O}}$ is defined relative to pyridine, a value of -11.93 kcal/mol from Arnett et al. (1977) was added.

**560)** Due to an apparently incorrect definition of the Henry's law constant by Andon et al. (1954),



Staudinger and Roberts (2001) quote incorrect values from that paper.

**561)** The data from Wieland et al. (2015) were fitted to the 3-parameter equation: $H_s^{cp} = \exp(-12.48322 + 7833.96799/T - 2.23379\ln(T))$ mol/(m$^3$ Pa) with $T$ in K.

**562)** Value given here as quoted by Feigenbrugel and Le Calvé (2021).

**563)** Value calculated from the solubility of $9.4 \times 10^{-3}$ mol/L and the vapor pressure of 0.255 mmHg, as shown on pages 7142-7143 of Arnett and Chawla (1979). It is inconsistent with the entry in Table IV of that paper.

**564)** Value given here as quoted by Ma et al. (2010a).

**565)** Nguyen (2013) refer to Kim et al. (2008) as the source but this value cannot be found there.

**566)** Values at 298 K in Table C2 and C5 of Brockbank (2013) are inconsistent, with 7 % difference.

**567)** Value given here as quoted by Goodarzi et al. (2010).

**568)** Goodarzi et al. (2010) compared several QSPR methods and found that the Levenberg-Marquardt algorithm with Bayesian regularization produces the best results. Values obtained with other methods can be found in their supplement.

**569)** Value from the validation set.

**570)** At pH 5.

**571)** Value from the test set.

**572)** At pH 10.

**573)** At pH 9.

**574)** At pH 5.2.

**575)** At pH 7.4.

**576)** At pH 9.3.

**577)** At pH 4.

**578)** Kames and Schurath (1992) were unable to assign the values to the isomers.

**579)** Ebert et al. (2023) present "curated experimental" Henry's law constants from the literature but do not provide any references. It is only mentioned that the value is from a single original paper and measured directly.

**580)** The same data were also published in Fischer and Ballschmiter (1998a).

**581)** The formula of 1,3-propanediol dinitrate is incorrectly given as "O$_2$NO$_2$CH$_2$CH$_2$CH$_2$ONO$_2$" by Burkholder et al. (2019).

**582)** The formula of 1,3-propanediol dinitrate is incorrectly given as "O$_2$NO$_2$CH$_2$CH$_2$CH$_2$ONO$_2$" by Burkholder et al. (2015).

**583)** Comparing the value with that from the cited publication (Kames and Schurath, 1995), it can be seen that the unit and the temperature listed in Table 3 of Warneck et al. (1996) are incorrect.

**584)** The data from Kames and Schurath (1995) for peroxyacetyl nitrate are incorrectly cited by Schurath et al. (1996).

**585)** The data from Kames and Schurath (1995) for peroxypropionyl nitrate are incorrectly cited by Schurath et al. (1996).

**586)** The data from Kames and Schurath (1995) for peroxy-$n$-butyl nitrate are incorrectly cited by Schurath et al. (1996).

**587)** The data from Kames and Schurath (1995) for peroxymethacryloyl nitrate are incorrectly cited by Schurath et al. (1996).

**588)** The data from Kames and Schurath (1995) for peroxy-$i$-butyl nitrate are incorrectly cited by Schurath et al. (1996).

**589)** Estimate based on Raventos-Duran et al. (2010).

**590)** Effective value at pH 3.

**591)** The value at $T^\ominus$ is the intrinsic Henry's law constant but the temperature dependence refers to the effective Henry's law constant at pH 3.08.

**592)** Burkholder et al. (2019) refer to Borduas et al. (2016) but the quoted value cannot be found there.

**593)** The values for nitroethane in Tabs. VI and VIII of Friant and Suffet (1979) differ by a factor of 10. Apparently, the value in Tab. VIII is wrong.

**594)** The data listed in Tabs. 2 and 3 of Dewulf et al. (1999) are inconsistent, with 27 % difference.

**595)** Mackay et al. (2006d) list two values for dinoseb which differ by a factor of 1000. It is unclear which number is correct (if any) and the data are not reproduced here.

**596)** Values at 298 K in Table C2 and C5 of Brockbank (2013) are inconsistent, with 6 % difference.

**597)** The data from Glew and Moelwyn-Hughes (1953) were fitted to the 3-parameter equation: $H_s^{cp} = \exp(-135.82151 + 7593.40134/T + 18.05983\ln(T))$ mol/(m$^3$ Pa) with $T$ in K.

**598)** The data from Maaßen (1995) were fitted to the 3-parameter equation: $H_s^{cp} = \exp(-163.70243 + 8973.31702/T + 22.17142\ln(T))$ mol/(m$^3$ Pa) with $T$ in K.

**599)** The data from Reichl (1995) were fitted to the 3-parameter equation: $H_s^{cp} = \exp(-142.70480 + 8025.53525/T + 19.04459\ln(T))$ mol/(m$^3$ Pa) with $T$ in K.





**600)** The data from Zheng et al. (1997) were fitted to the 3-parameter equation: $H_s^{cp} =$ $\exp(-190.61883 + 10088.26604/T + 25.94088 \ln(T))$ mol/(m$^3$ Pa) with $T$ in K.

**601)** The data from Maaßen (1995) were fitted to the 3-parameter equation: $H_s^{cp} =$ $\exp(-177.44258 + 9554.69077/T + 23.94054 \ln(T))$ mol/(m$^3$ Pa) with $T$ in K.

**602)** The data from Reichl (1995) were fitted to the 3-parameter equation: $H_s^{cp} =$ $\exp(-350.64777 + 16708.21486/T + 49.40261 \ln(T))$ mol/(m$^3$ Pa) with $T$ in K.

**603)** The data from Scharlin and Battino (1995) were fitted to the 3-parameter equation: $H_s^{cp} =$ $\exp(-552.21779 + 25529.81258/T + 79.59510 \ln(T))$ mol/(m$^3$ Pa) with $T$ in K.

**604)** The data from Scharlin and Battino (1994) were fitted to the 3-parameter equation: $H_s^{cp} =$ $\exp(-552.21779 + 25529.81258/T + 79.59510 \ln(T))$ mol/(m$^3$ Pa) with $T$ in K.

**605)** The data from Wen and Muccitelli (1979) were fitted to the 3-parameter equation: $H_s^{cp} =$ $\exp(-356.93310 + 16943.80173/T + 50.37092 \ln(T))$ mol/(m$^3$ Pa) with $T$ in K.

**606)** The data from Ashton et al. (1968) were fitted to the 3-parameter equation: $H_s^{cp} =$ $\exp(-320.94892 + 15261.58540/T + 45.04995 \ln(T))$ mol/(m$^3$ Pa) with $T$ in K.

**607)** The data from Morrison and Johnstone (1954) were fitted to the 3-parameter equation: $H_s^{cp} =$ $\exp(-174.44927 + 8434.85415/T + 23.34667 \ln(T))$ mol/(m$^3$ Pa) with $T$ in K.

**608)** The H298 and A,B data listed in Table 5-7 of Burkholder et al. (2019) are inconsistent, with 8 % difference.

**609)** The H298 and A,B data listed in Table 5-7 of Burkholder et al. (2015) are inconsistent, with 8 % difference.

**610)** The data from Zheng et al. (1997) were fitted to the 3-parameter equation: $H_s^{cp} =$ $\exp(-203.78636 + 11097.46295/T + 27.89781 \ln(T))$ mol/(m$^3$ Pa) with $T$ in K.

**611)** The data from Maaßen (1995) were fitted to the 3-parameter equation: $H_s^{cp} =$ $\exp(-184.82864 + 10260.68840/T + 25.06659 \ln(T))$ mol/(m$^3$ Pa) with $T$ in K.

**612)** The data from Reichl (1995) were fitted to the 3-parameter equation: $H_s^{cp} =$ $\exp(-175.64793 + 9805.36391/T + 23.71997 \ln(T))$ mol/(m$^3$ Pa) with $T$ in K.

**613)** The data from Zheng et al. (1997) were fitted to the 3-parameter equation: $H_s^{cp} =$ $\exp(-244.13803 + 12963.44791/T + 33.68869 \ln(T))$ mol/(m$^3$ Pa) with $T$ in K.

**614)** The data from Maaßen (1995) were fitted to the 3-parameter equation: $H_s^{cp} =$ $\exp(-225.56576 + 12186.49271/T + 30.88527 \ln(T))$ mol/(m$^3$ Pa) with $T$ in K.

**615)** The data from Reichl (1995) were fitted to the 3-parameter equation: $H_s^{cp} =$ $\exp(-208.89051 + 11387.65726/T + 28.42219 \ln(T))$ mol/(m$^3$ Pa) with $T$ in K.

**616)** The data from Chang and Criddle (1995) were fitted to the 3-parameter equation: $H_s^{cp} = \exp(-1003.84803 + 45506.40253/T + 147.89569 \ln(T))$ mol/(m$^3$ Pa) with $T$ in K.

**617)** The data from Reichl (1995) were fitted to the 3-parameter equation: $H_s^{cp} =$ $\exp(-164.25882 + 9381.26592/T + 21.50848 \ln(T))$ mol/(m$^3$ Pa) with $T$ in K.

**618)** The data from Wen and Muccitelli (1979) were fitted to the 3-parameter equation: $H_s^{cp} =$ $\exp(-499.57565 + 23563.38593/T + 71.28478 \ln(T))$ mol/(m$^3$ Pa) with $T$ in K.

**619)** Value at $T$ = 287 K.

**620)** The data from Reichl (1995) were fitted to the 3-parameter equation: $H_s^{cp} =$ $\exp(-78.74672 + 5836.90728/T + 8.41930 \ln(T))$ mol/(m$^3$ Pa) with $T$ in K.

**621)** The data from Clever et al. (2005) were fitted to the 3-parameter equation: $H_s^{cp} =$ $\exp(-588.11467 + 28143.61522/T + 84.26598 \ln(T))$ mol/(m$^3$ Pa) with $T$ in K.

**622)** In their Table 13, Clever et al. (2005) list Ostwald coefficients that are probably incorrect by a factor of 100. Therefore, these values are not used. Instead, $H_s$ is calculated using the mol fraction $x_1$ from the same table.

**623)** The data from Scharlin and Battino (1994) were fitted to the 3-parameter equation: $H_s^{cp} =$ $\exp(-630.69809 + 30309.09484/T + 90.46889 \ln(T))$ mol/(m$^3$ Pa) with $T$ in K.

**624)** The data from Wen and Muccitelli (1979) were fitted to the 3-parameter equation: $H_s^{cp} =$ $\exp(-673.45393 + 31915.35190/T + 97.01332 \ln(T))$ mol/(m$^3$ Pa) with $T$ in K.

**625)** The data from Maaßen (1995) were fitted to the 3-parameter equation: $H_s^{cp} =$ $\exp(-197.14327 + 10473.25304/T + 26.34780 \ln(T))$ mol/(m$^3$ Pa) with $T$ in K.

**626)** Calculated using the COSMO-RS method.



**627)** Value given here as quoted by Kanakidou et al. (1995).

**628)** Comparing with Abraham et al. (1994a), it seems that the compound called "trifluoroethanol" by Goss (2005) refers to 2,2,2-trifluoroethanol.

**629)** Comparing with Abraham et al. (1994a), it seems that the compound called "hexafluoropropanol" by Goss (2005) refers to 1,1,1,3,3,3-hexafluoro-2-propanol.

**630)** Value at $T = 284$ K.

**631)** Measured in aqueous hydrochloric acid and extrapolated to pure water as the solvent at 25 °C. Measurements were also made at other temperatures, however, those were not extrapolated to pure water as the solvent.

**632)** Calculated using the EPI Suite Bond estimation method.

**633)** Calculated using the new SPARC method, see Arp et al. (2006) for details.

**634)** Calculated using the COSMOtherm method, see Arp et al. (2006) for details.

**635)** A refit yields: A = -18.99, B = 5493, and H(298 K) = 0.57 M/atm (R. E. Huie, pers. comm. 2021).

**636)** A refit yields: A = -18.99, B = 5493, and H(298 K) = 0.57 M/atm (R. E. Huie, pers. comm. 2021).

**637)** A refit yields: A = -21.67, B = 5776, and H(298 K) = 0.10 M/atm (R. E. Huie, pers. comm. 2021).

**638)** The H298 and A,B data listed in Table 5-4 of Burkholder et al. (2019) are inconsistent, with 29 % difference.

**639)** A refit yields: A = -21.67, B = 5776, and H(298 K) = 0.10 M/atm (R. E. Huie, pers. comm. 2021).

**640)** The H298 and A,B data listed in Table 5-4 of Burkholder et al. (2015) are inconsistent, with 29 % difference.

**641)** The H298 and A,B data listed in Table 5.4 of Sander et al. (2011) are inconsistent, with 29 % difference.

**642)** The data from Clever et al. (2005) were fitted to the 3-parameter equation: $H_s^{cp} = \exp(289.52696 - 11352.27202/T - 46.16631\ln(T))$ mol/(m$^3$ Pa) with $T$ in K.

**643)** The Ostwald coefficient given by Clever et al. (2005) at 313.2 K is probably incorrect. Therefore, the Ostwald coefficients are not used. Instead, $H_s$ is calculated using the mol fraction $x_1$ from the same table.

**644)** Extrapolated based on number of carbons.

**645)** Measured with the wetted-wall column at room temperature.

**646)** The H298 and A,B data listed in Table 5.4 of Sander et al. (2011) are inconsistent, with 9 % difference.

**647)** The H298 and A,B data listed in Table 5.4 of Sander et al. (2006) are inconsistent, with 9 % difference.

**648)** The data from Reichl (1995) were fitted to the 3-parameter equation: $H_s^{cp} = \exp(-251.05500 + 13259.10200/T + 35.01685\ln(T))$ mol/(m$^3$ Pa) with $T$ in K.

**649)** The same data were also published in McConnell et al. (1975).

**650)** The data from Glew and Moelwyn-Hughes (1953) were fitted to the 3-parameter equation: $H_s^{cp} = \exp(-171.13914 + 9743.00524/T + 23.09616\ln(T))$ mol/(m$^3$ Pa) with $T$ in K.

**651)** Values at different temperatures are from different sources. Thus a temperature dependence was not calculated.

**652)** Chiang et al. (1998) show vinyl chloride in their Table 2 but most probably they meant to refer to dichloromethane instead.

**653)** The data from Wright et al. (1992) were fitted to the 3-parameter equation: $H_s^{cp} = \exp(-444.17924 + 22456.73010/T + 63.76504\ln(T))$ mol/(m$^3$ Pa) with $T$ in K.

**654)** Values at 298 K in Table C2 and C5 of Brockbank (2013) are inconsistent, with 7 % difference.

**655)** The data from Görgényi et al. (2002) were fitted to the 3-parameter equation: $H_s^{cp} = \exp(-378.59438 + 20174.67146/T + 53.50889\ln(T))$ mol/(m$^3$ Pa) with $T$ in K.

**656)** The data from Moore et al. (1995) were fitted to the 3-parameter equation: $H_s^{cp} = \exp(32.52949 + 1878.33965/T - 7.88669\ln(T))$ mol/(m$^3$ Pa) with $T$ in K.

**657)** The data from Wright et al. (1992) were fitted to the 3-parameter equation: $H_s^{cp} = \exp(-109.09283 + 8000.75665/T + 13.39152\ln(T))$ mol/(m$^3$ Pa) with $T$ in K.

**658)** Probably an interpolation of the data from Balls (1980).

**659)** Values at 298 K in Table C2 and C5 of Brockbank (2013) are inconsistent, with 5 % difference.

**660)** The data from Bullister and Wisegarver (1998) were fitted to the 3-parameter equation: $H_s^{cp} = \exp(-704.15798 + 34144.64622/T + 102.06046\ln(T))$ mol/(m$^3$ Pa) with $T$ in K.

**661)** The data from Wright et al. (1992) were fitted to the 3-parameter equation: $H_s^{cp} = \exp(-426.63883 + 22457.44484/T + 60.22986\ln(T))$ mol/(m$^3$ Pa) with $T$ in K.





**662)** The data from Maaßen (1995) were fitted to the 3-parameter equation: $H_s^{cp} = \exp(-221.58683 + 12291.19608/T + 30.42274 \ln(T))$ mol/(m$^3$ Pa) with $T$ in K.

**663)** The data from Reichl (1995) were fitted to the 3-parameter equation: $H_s^{cp} = \exp(-233.85465 + 12927.81251/T + 32.20905 \ln(T))$ mol/(m$^3$ Pa) with $T$ in K.

**664)** The data from Görgényi et al. (2002) were fitted to the 3-parameter equation: $H_s^{cp} = \exp(-372.18420 + 19566.35271/T + 52.67600 \ln(T))$ mol/(m$^3$ Pa) with $T$ in K.

**665)** The data from Wright et al. (1992) were fitted to the 3-parameter equation: $H_s^{cp} = \exp(-1295.59488 + 61538.96732/T + 190.02999 \ln(T))$ mol/(m$^3$ Pa) with $T$ in K.

**666)** The data from Wright et al. (1992) were fitted to the 3-parameter equation: $H_s^{cp} = \exp(-309.75754 + 17275.24359/T + 43.35857 \ln(T))$ mol/(m$^3$ Pa) with $T$ in K.

**667)** The data from Robbins et al. (1993) were fitted to the 3-parameter equation: $H_s^{cp} = \exp(313.50875 - 12121.71831/T - 49.20602 \ln(T))$ mol/(m$^3$ Pa) with $T$ in K.

**668)** The data from Wright et al. (1992) were fitted to the 3-parameter equation: $H_s^{cp} = \exp(255.46482 - 8896.18926/T - 40.90189 \ln(T))$ mol/(m$^3$ Pa) with $T$ in K.

**669)** Values at 298 K in Table C2 and C5 of Brockbank (2013) are inconsistent, with 5 % difference.

**670)** The data from Schwardt et al. (2021) were fitted to the 3-parameter equation: $H_s^{cp} = \exp(200.57633 - 7128.32092/T - 31.87111 \ln(T))$ mol/(m$^3$ Pa) with $T$ in K.

**671)** The data from Wright et al. (1992) were fitted to the 3-parameter equation: $H_s^{cp} = \exp(-304.31063 + 17046.46392/T + 42.59182 \ln(T))$ mol/(m$^3$ Pa) with $T$ in K.

**672)** The data from Schwardt et al. (2021) were fitted to the 3-parameter equation: $H_s^{cp} = \exp(-1784.40256 + 88283.19114/T + 260.26556 \ln(T))$ mol/(m$^3$ Pa) with $T$ in K.

**673)** The data from Wright et al. (1992) were fitted to the 3-parameter equation: $H_s^{cp} = \exp(608.52671 - 23622.70039/T - 93.86675 \ln(T))$ mol/(m$^3$ Pa) with $T$ in K.

**674)** The value for $A$ in the table of Kondoh and Nakajima (1997) is incorrect. Recalculating the regression, it can be seen that it should be 13.95 and not 1.395.

**675)** The data from Wright et al. (1992) were fitted to the 3-parameter equation: $H_s^{cp} = \exp(2638.58362 - 114985.14319/T - 396.08684 \ln(T))$ mol/(m$^3$ Pa) with $T$ in K.

**676)** The H298 and A,B data listed in Table 5-4 of Burkholder et al. (2019) are inconsistent, with 9 % difference.

**677)** The data from Wright et al. (1992) were fitted to the 3-parameter equation: $H_s^{cp} = \exp(-573.71583 + 28877.33987/T + 82.70652 \ln(T))$ mol/(m$^3$ Pa) with $T$ in K.

**678)** The data from Sarraute et al. (2006) were fitted to the 3-parameter equation: $H_s^{cp} = \exp(-503.53929 + 28223.72051/T + 70.89539 \ln(T))$ mol/(m$^3$ Pa) with $T$ in K.

**679)** As explained by Miller and Stuart (2003), the measurements were performed at 296 K.

**680)** Value for $T = 295\ldots298$ K.

**681)** Value for $T = 293\ldots298$ K.

**682)** The data from Schwardt et al. (2021) were fitted to the 3-parameter equation: $H_s^{cp} = \exp(-20.66741 + 2604.13624/T + 0.71646 \ln(T))$ mol/(m$^3$ Pa) with $T$ in K.

**683)** Mackay et al. (2006b) list a vapor pressure $p$, a solubility $c$, and a Henry's law constant calculated as $p/c$. However, the data are internally inconsistent and deviate by more than 10 %.

**684)** Values at 298 K in Table C2 and C5 of Brockbank (2013) are inconsistent, with 6 % difference.

**685)** The data from Schwardt et al. (2021) were fitted to the 3-parameter equation: $H_s^{cp} = \exp(-214.72727 + 12076.60512/T + 29.20360 \ln(T))$ mol/(m$^3$ Pa) with $T$ in K.

**686)** A typo in Ashworth et al. (1988) has been corrected, the correct value was presented by Howe et al. (1987).

**687)** The data from Wright et al. (1992) were fitted to the 3-parameter equation: $H_s^{cp} = \exp(-332.68901 + 17925.88529/T + 46.77838 \ln(T))$ mol/(m$^3$ Pa) with $T$ in K.

**688)** The data from Schwardt et al. (2021) were fitted to the 3-parameter equation: $H_s^{cp} = \exp(-6.68864 + 2211.35284/T - 1.35565 \ln(T))$ mol/(m$^3$ Pa) with $T$ in K.

**689)** The data from Khalfaoui and Newsham (1994b) were fitted to the 3-parameter equation: $H_s^{cp} = \exp(-593.56757 + 30300.79738/T + 85.11672 \ln(T))$ mol/(m$^3$ Pa) with $T$ in K.

**690)** The data from Wright et al. (1992) were fitted to the 3-parameter equation: $H_s^{cp} = \exp(-294.54970 + 16409.35487/T + 40.82700 \ln(T))$ mol/(m$^3$ Pa) with $T$ in K.



**691)** The data from Cooling et al. (1992) were fitted to the 3-parameter equation: $H_s^{cp} = \exp(-511.78322 + 26710.11950/T + 72.88403 \ln(T))$ mol/(m³ Pa) with $T$ in K.

**692)** The data from Schwardt et al. (2021) were fitted to the 3-parameter equation: $H_s^{cp} = \exp(-265.05147 + 15058.79780/T + 36.44507 \ln(T))$ mol/(m³ Pa) with $T$ in K.

**693)** The data from Görgényi et al. (2002) were fitted to the 3-parameter equation: $H_s^{cp} = \exp(-480.92432 + 24776.46284/T + 68.60174 \ln(T))$ mol/(m³ Pa) with $T$ in K.

**694)** The data from Knauss et al. (2000) were fitted to the 3-parameter equation: $H_s^{cp} = \exp(-389.28726 + 21123.08804/T + 54.69871 \ln(T))$ mol/(m³ Pa) with $T$ in K.

**695)** The data from Khalfaoui and Newsham (1994b) were fitted to the 3-parameter equation: $H_s^{cp} = \exp(-511.93773 + 26713.30359/T + 72.90551 \ln(T))$ mol/(m³ Pa) with $T$ in K.

**696)** The data from Robbins et al. (1993) were fitted to the 3-parameter equation: $H_s^{cp} = \exp(176.56015 - 5511.47473/T - 28.96682 \ln(T))$ mol/(m³ Pa) with $T$ in K.

**697)** The data from Wright et al. (1992) were fitted to the 3-parameter equation: $H_s^{cp} = \exp(681.41357 - 27448.54898/T - 104.63745 \ln(T))$ mol/(m³ Pa) with $T$ in K.

**698)** The data from Cooling et al. (1992) were fitted to the 3-parameter equation: $H_s^{cp} = \exp(-574.03630 + 29404.80442/T + 82.22224 \ln(T))$ mol/(m³ Pa) with $T$ in K.

**699)** The data from Schwardt et al. (2021) were fitted to the 3-parameter equation: $H_s^{cp} = \exp(-330.94781 + 18207.73829/T + 46.05991 \ln(T))$ mol/(m³ Pa) with $T$ in K.

**700)** The data from Knauss et al. (2000) were fitted to the 3-parameter equation: $H_s^{cp} = \exp(-281.09217 + 15955.08953/T + 38.60107 \ln(T))$ mol/(m³ Pa) with $T$ in K.

**701)** The data from Robbins et al. (1993) were fitted to the 3-parameter equation: $H_s^{cp} = \exp(-1145.60543 + 55089.35358/T + 167.32916 \ln(T))$ mol/(m³ Pa) with $T$ in K.

**702)** The data from Wright et al. (1992) were fitted to the 3-parameter equation: $H_s^{cp} = \exp(-224.85290 + 13463.70772/T + 30.65123 \ln(T))$ mol/(m³ Pa) with $T$ in K.

**703)** Henry's law constants were evaluated using data from Florida sandy field soil.

**704)** According to Thomas et al. (2006), theoretical Henry's law constants were calculated using the "normal boiling point, the critical temperature, and the enthalpy of volatilization at the normal boiling point".

**705)** Haynes (2014) refer to Mackay and Shiu (1981) but that article lists this value for 1-chloro-2-methylpropane (the saturated compound), not for 1-chloro-2-methylpropene.

**706)** The regression parameters for chlorobenzene in Tab. 1 of Schwardt et al. (2021) are wrong. Corrected values from Schwardt et al. (2022) are used here.

**707)** The data from Schwardt et al. (2021) were fitted to the 3-parameter equation: $H_s^{cp} = \exp(-266.69788 + 14811.78372/T + 37.00246 \ln(T))$ mol/(m³ Pa) with $T$ in K.

**708)** The data from Khalfaoui and Newsham (1994b) were fitted to the 3-parameter equation: $H_s^{cp} = \exp(-757.46460 + 35956.18738/T + 110.75693 \ln(T))$ mol/(m³ Pa) with $T$ in K.

**709)** The data from Cooling et al. (1992) were fitted to the 3-parameter equation: $H_s^{cp} = \exp(-820.00716 + 38880.20610/T + 120.01460 \ln(T))$ mol/(m³ Pa) with $T$ in K.

**710)** Values at 298 K in Table C2 and C5 of Brockbank (2013) are inconsistent, with 8 % difference.

**711)** Erratum for page 344 of Fogg and Sangster (2003): Their reference [89] does not contain 1,2-dichlorobenzene.

**712)** The data listed in Tabs. 2 and 3 of Dewulf et al. (1999) are inconsistent, with 7 % difference.

**713)** The data listed in Tabs. 2 and 3 of Dewulf et al. (1999) are inconsistent, with 7 % difference.

**714)** Rumble (2021) refers to Oliver (1985) as the source but this value cannot be found there.

**715)** Value for $T$ = 298...303 K.

**716)** Ebert et al. (2023) present "curated experimental" Henry's law constants from the literature but do not provide any references. It is only mentioned that the value is from multiple sources and obtained from the experimental octanol/water partition coefficient and the octanol/air partition coefficient.

**717)** Ebert et al. (2023) present "curated experimental" Henry's law constants from the literature but do not provide any references. It is only mentioned that the value is from a single database or data collection and obtained from the experimental octanol/water partition coefficient and the octanol/air partition coefficient.

**718)** Odabasi and Adali (2016) provide the parameters $A$ and $B$ for an equation to calculate temperature-dependent Henry's law constants. Values



calculated with this equation at 298 K are slightly different than those measured at 298 K and given as H in their table 1. Here, the values H and B are used.

**719)** Modified gas-stripping method (MGSM), see Lau et al. (2006) for details.

**720)** Integrated gas-stripping method (IGSM), see Lau et al. (2006) for details.

**721)** Calculated with the principal component regression (PCR) method, see Fang Lee (2007) for details.

**722)** Calculated with the partial least-square regression (PLSR) method, see Fang Lee (2007) for details.

**723)** The same data were also published in Dunnivant et al. (1988).

**724)** Value given here as quoted by Dunnivant et al. (1988).

**725)** Calculated using the EPICS method.

**726)** Calculated using the "Direct" method.

**727)** Westcott et al. (1981) give a range of $1.9{\times}10^{-2}$ mol/(m$^3$ Pa) $< H_{\rm s}^{cp} < 3.2{\times}10^{-2}$ mol/(m$^3$ Pa).

**728)** Westcott et al. (1981) give a range of $2.8{\times}10^{-2}$ mol/(m$^3$ Pa) $< H_{\rm s}^{cp} < 9.0{\times}10^{-2}$ mol/(m$^3$ Pa).

**729)** Erratum for page 350 of Fogg and Sangster (2003): The equation describing the recommended temperature-dependent data appears to be incorrect and is not used here.

**730)** Value at pH = 4.

**731)** When comparing $H$ in Table 4 with $K_{\rm gw}$ in Table 5 of Pfeifer et al. (2001), it can be seen that the values refer to $K_{\rm gw} \times 100$ and not $K_{\rm gw}/100$.

**732)** Measured at pH 1.

**733)** The same data were also published in Brandsch et al. (1993).

**734)** Erratum for page 376 of Fogg and Sangster (2003): Data from Santl et al. (1994) are cited incorrectly, it should be 3.64, not 3.84.

**735)** Values at 298 K in Table C2 and C5 of Brockbank (2013) are inconsistent, with 8 % difference.

**736)** Although pronamide and propyzamide are the same species, Mackay et al. (2006d) list two different values for them. It is unclear which number is correct (if any) and the data are not reproduced here.

**737)** The value at 20 °C was calculated from published values of vapor pressure and water solubility. Data between 25 °C and 40 °C were calculated from the measured evaporation rate.

**738)** At pH 5.4.

**739)** Ebert et al. (2023) present "curated experimental" Henry's law constants from the literature but do not provide any references. It is only mentioned that the value is from multiple sources and measured directly.

**740)** The data from Zheng et al. (1997) were fitted to the 3-parameter equation: $H_{\rm s}^{cp} = \exp(-206.94328 + 11372.60160/T + 28.22232\ln(T))$ mol/(m$^3$ Pa) with $T$ in K.

**741)** The data from Maaßen (1995) were fitted to the 3-parameter equation: $H_{\rm s}^{cp} = \exp(-200.57402 + 11192.93914/T + 27.21798\ln(T))$ mol/(m$^3$ Pa) with $T$ in K.

**742)** The data from Reichl (1995) were fitted to the 3-parameter equation: $H_{\rm s}^{cp} = \exp(-208.06388 + 11491.48483/T + 28.35421\ln(T))$ mol/(m$^3$ Pa) with $T$ in K.

**743)** The data from Chang and Criddle (1995) were fitted to the 3-parameter equation: $H_{\rm s}^{cp} = \exp(-1756.79407 + 80807.02552/T + 259.24906\ln(T))$ mol/(m$^3$ Pa) with $T$ in K.

**744)** The data from McLinden (1989) were fitted to the 3-parameter equation: $H_{\rm s}^{cp} = \exp(-387.81156 + 19950.78638/T + 54.91348\ln(T))$ mol/(m$^3$ Pa) with $T$ in K.

**745)** The temperature dependence was recalculated from the data on p. 20 of McLinden (1989).

**746)** The data from McLinden (1989) for HCFC-22 are incorrectly cited by Kanakidou et al. (1995).

**747)** The H298 and A,B data listed in Table 5-4 of Burkholder et al. (2019) are inconsistent, with 5 % difference.

**748)** The H298 and A,B data listed in Table 5-4 of Burkholder et al. (2015) are inconsistent, with 5 % difference.

**749)** The H298 and A,B data listed in Table 5.4 of Sander et al. (2011) are inconsistent, with 5 % difference.

**750)** The data from Reichl (1995) were fitted to the 3-parameter equation: $H_{\rm s}^{cp} = \exp(100.23590 - 3339.68982/T - 17.66849\ln(T))$ mol/(m$^3$ Pa) with $T$ in K.

**751)** The data from Scharlin and Battino (1995) were fitted to the 3-parameter equation: $H_{\rm s}^{cp} = \exp(-291.40685 + 14224.53456/T + 40.73325\ln(T))$ mol/(m$^3$ Pa) with $T$ in K.

**752)** The data from Scharlin and Battino (1994) were fitted to the 3-parameter equation: $H_{\rm s}^{cp} = \exp(-291.40685 + 14224.53456/T + 40.73325\ln(T))$ mol/(m$^3$ Pa) with $T$ in K.

**753)** The data from Reichl (1995) were fitted to the 3-parameter equation: $H_{\rm s}^{cp} = \exp(-147.53824 + 8643.05363/T + 18.97752\ln(T))$ mol/(m$^3$ Pa) with $T$ in K.

**754)** The data from Scharlin and Battino (1995) were fitted to the 3-parameter equation: $H_{\rm s}^{cp} =$



$\exp(-211.99699 + 11400.41036/T + 28.66283\ln(T))$ mol/(m$^3$ Pa) with $T$ in K.

**755)** The data from Scharlin and Battino (1994) were fitted to the 3-parameter equation: $H_s^{cp} =$ $\exp(-211.99699 + 11400.41036/T + 28.66283\ln(T))$ mol/(m$^3$ Pa) with $T$ in K.

**756)** The data from Maaßen (1995) were fitted to the 3-parameter equation: $H_s^{cp} =$ $\exp(-278.68448 + 15169.41095/T + 38.36974\ln(T))$ mol/(m$^3$ Pa) with $T$ in K.

**757)** The data from Reichl (1995) were fitted to the 3-parameter equation: $H_s^{cp} =$ $\exp(-129.78084 + 8533.77911/T + 16.20428\ln(T))$ mol/(m$^3$ Pa) with $T$ in K.

**758)** The data from Bu and Warner (1995) were fitted to the 3-parameter equation: $H_s^{cp} =$ $\exp(-415.59157 + 21411.24346/T + 58.50528\ln(T))$ mol/(m$^3$ Pa) with $T$ in K.

**759)** The data from Reichl (1995) were fitted to the 3-parameter equation: $H_s^{cp} = \exp(0.13353 + 5070.08549/T - 4.84639\ln(T))$ mol/(m$^3$ Pa) with $T$ in K.

**760)** The data from Reichl (1995) were fitted to the 3-parameter equation: $H_s^{cp} =$ $\exp(-113.07654 + 6884.36758/T + 13.75470\ln(T))$ mol/(m$^3$ Pa) with $T$ in K.

**761)** The data from Reichl (1995) were fitted to the 3-parameter equation: $H_s^{cp} =$ $\exp(-149.62353 + 7869.46528/T + 19.40044\ln(T))$ mol/(m$^3$ Pa) with $T$ in K.

**762)** The data from Chang and Criddle (1995) were fitted to the 3-parameter equation: $H_s^{cp} =$ $\exp(-402.28495 + 20229.16189/T + 57.28419\ln(T))$ mol/(m$^3$ Pa) with $T$ in K.

**763)** The data from Maaßen (1995) were fitted to the 3-parameter equation: $H_s^{cp} =$ $\exp(-277.88370 + 14905.51805/T + 38.38688\ln(T))$ mol/(m$^3$ Pa) with $T$ in K.

**764)** The data from Reichl (1995) were fitted to the 3-parameter equation: $H_s^{cp} =$ $\exp(-195.56650 + 11207.08869/T + 26.12575\ln(T))$ mol/(m$^3$ Pa) with $T$ in K.

**765)** The data from Maaßen (1995) were fitted to the 3-parameter equation: $H_s^{cp} =$ $\exp(-249.15404 + 13774.89590/T + 34.23234\ln(T))$ mol/(m$^3$ Pa) with $T$ in K.

**766)** The data from Reichl (1995) were fitted to the 3-parameter equation: $H_s^{cp} =$ $\exp(-210.55601 + 11968.42846/T + 28.54087\ln(T))$ mol/(m$^3$ Pa) with $T$ in K.

**767)** The data from Maaßen (1995) were fitted to the 3-parameter equation: $H_s^{cp} =$ $\exp(-155.04312 + 9704.04801/T + 20.06575\ln(T))$ mol/(m$^3$ Pa) with $T$ in K.

**768)** The data from Maaßen (1995) were fitted to the 3-parameter equation: $H_s^{cp} =$ $\exp(-270.78344 + 14413.03953/T + 37.48366\ln(T))$ mol/(m$^3$ Pa) with $T$ in K.

**769)** The data from Reichl (1995) were fitted to the 3-parameter equation: $H_s^{cp} =$ $\exp(-184.96240 + 10541.13831/T + 24.70437\ln(T))$ mol/(m$^3$ Pa) with $T$ in K.

**770)** The data from Chang and Criddle (1995) were fitted to the 3-parameter equation: $H_s^{cp} =$ $\exp(-190.58060 + 10602.65774/T + 25.66197\ln(T))$ mol/(m$^3$ Pa) with $T$ in K.

**771)** The data from Maaßen (1995) were fitted to the 3-parameter equation: $H_s^{cp} =$ $\exp(-237.50724 + 13032.41274/T + 32.48569\ln(T))$ mol/(m$^3$ Pa) with $T$ in K.

**772)** The data from Reichl (1995) were fitted to the 3-parameter equation: $H_s^{cp} =$ $\exp(-200.95912 + 11406.81841/T + 27.03092\ln(T))$ mol/(m$^3$ Pa) with $T$ in K.

**773)** The data from Smith et al. (1981b) were fitted to the 3-parameter equation: $H_s^{cp} =$ $\exp(678.00770 - 27346.39638/T - 103.92351\ln(T))$ mol/(m$^3$ Pa) with $T$ in K.

**774)** Kanakidou et al. (1995) assume $H_s(\mathrm{CClF_2OONO_2})$ $= H_s(\mathrm{PAN})$.

**775)** The H298 and A,B data listed in Table 5-4 of Burkholder et al. (2019) are inconsistent, with 11 % difference.

**776)** The data from De Bruyn and Saltzman (1997) were fitted to the 3-parameter equation: $H_s^{cp} =$ $\exp(-521.17646 + 25057.64644/T + 75.60914\ln(T))$ mol/(m$^3$ Pa) with $T$ in K.

**777)** The data from Glew and Moelwyn-Hughes (1953) were fitted to the 3-parameter equation: $H_s^{cp} =$ $\exp(-184.73597 + 10636.09284/T + 25.03175\ln(T))$ mol/(m$^3$ Pa) with $T$ in K.

**778)** Values at 298 K in Table C2 and C5 of Brockbank (2013) are inconsistent, with 6 % difference.

**779)** The data from Moore et al. (1995) were fitted to the 3-parameter equation: $H_s^{cp} =$ $\exp(-395.20167 + 20638.03484/T + 56.40082\ln(T))$ mol/(m$^3$ Pa) with $T$ in K.

**780)** The data from Wright et al. (1992) were fitted to the 3-parameter equation: $H_s^{cp} =$ $\exp(-82.06673 + 6867.92071/T + 9.56720\ln(T))$ mol/(m$^3$ Pa) with $T$ in K.



**781)** The data from Moore et al. (1995) were fitted to the 3-parameter equation: $H_s^{cp} = \exp(-408.59491 + 21699.59623/T + 58.19801 \ln(T))$ mol/(m$^3$ Pa) with $T$ in K.

**782)** The data from Wright et al. (1992) were fitted to the 3-parameter equation: $H_s^{cp} = \exp(1124.79951 - 46767.40872/T - 170.54217 \ln(T))$ mol/(m$^3$ Pa) with $T$ in K.

**783)** Erratum for page 274 of Fogg and Sangster (2003): The value in the table is $k_H$, not $\ln k_H$.

**784)** Value at $T = 50$ K.

**785)** Rumble (2021) refers to Hiatt (2013) as the source but this value cannot be found there.

**786)** Haynes (2014) refer to Mackay et al. (1993) as the source but this value cannot be found there.

**787)** The data from Sarraute et al. (2006) were fitted to the 3-parameter equation: $H_s^{cp} = \exp(-370.06283 + 22192.71634/T + 51.12683 \ln(T))$ mol/(m$^3$ Pa) with $T$ in K.

**788)** Erratum for page 321 of Fogg and Sangster (2003): Data from Yates and Gan (1998) are cited with a typo. The value at 313.2 K should probably be $4.78 \times 10^{-6}$, not $4.78 \times 10^{-2}$.

**789)** Ebert et al. (2023) present "curated experimental" Henry's law constants from the literature but do not provide any references. It is only mentioned that the value is from a single original paper and obtained from the experimental octanol/water partition coefficient and the octanol/air partition coefficient.

**790)** Diaz et al. (2005) also cite a Henry's law constant from Pfeifer et al. (2001) even though this species is not mentioned there. There might be a mix up of the different haloanisoles.

**791)** Erratum for page 285 of Fogg and Sangster (2003): The data in their table look strange (9.70R) and are not used here.

**792)** The data from Moore et al. (1995) were fitted to the 3-parameter equation: $H_s^{cp} = \exp(-76.31131 + 7250.73360/T + 8.15388 \ln(T))$ mol/(m$^3$ Pa) with $T$ in K.

**793)** The data from Moore et al. (1995) were fitted to the 3-parameter equation: $H_s^{cp} = \exp(-265.18008 + 15516.80509/T + 36.54803 \ln(T))$ mol/(m$^3$ Pa) with $T$ in K.

**794)** The data from Moore et al. (1995) were fitted to the 3-parameter equation: $H_s^{cp} = \exp(-229.06923 + 13418.39257/T + 31.15669 \ln(T))$ mol/(m$^3$ Pa) with $T$ in K.

**795)** The data from Glew and Moelwyn-Hughes (1953) were fitted to the 3-parameter equation: $H_s^{cp} =$

**796)** $\exp(-384.31677 + 19391.25580/T + 54.93602 \ln(T))$ mol/(m$^3$ Pa) with $T$ in K.

**796)** The data from Moore et al. (1995) were fitted to the 3-parameter equation: $H_s^{cp} = \exp(-247.93525 + 14910.30572/T + 34.08071 \ln(T))$ mol/(m$^3$ Pa) with $T$ in K.

**797)** The regression given by Fogg and Sangster (2003) does not produce the data in their table. Thus the regression was recalculated.

**798)** The value listed as $A$ for iodobenzene is probably not $A$ but the Henry's law volatility constant $H_v^{px}$ at 298 K. For the value of $B$, a minus sign is probably missing.

**799)** The data from Moore et al. (1995) were fitted to the 3-parameter equation: $H_s^{cp} = \exp(-242.58767 + 14043.89458/T + 33.48497 \ln(T))$ mol/(m$^3$ Pa) with $T$ in K.

**800)** Karagodin-Doyennel et al. (2021) probably assume that $CH_2BrI$ has the same Henry's law constant as $CH_2ClI$.

**801)** The data from Haimi et al. (2006) were fitted to the 3-parameter equation: $H_s^{cp} = \exp(-266.45850 + 15036.99733/T + 36.80758 \ln(T))$ mol/(m$^3$ Pa) with $T$ in K.

**802)** Yaws et al. (2003) present Henry's law constants based on water solubility and vapor pressure. The water solubility is calculated using a correlation to the boiling point. For the vapor pressures, no references are provided.

**803)** The data from Zin et al. (2016) were fitted to the 3-parameter equation: $H_s^{cp} = \exp(-419.66332 + 22034.35758/T + 59.55571 \ln(T))$ mol/(m$^3$ Pa) with $T$ in K.

**804)** The data from Haimi et al. (2006) were fitted to the 3-parameter equation: $H_s^{cp} = \exp(-166.97891 + 10357.07398/T + 22.04420 \ln(T))$ mol/(m$^3$ Pa) with $T$ in K.

**805)** The data from Haimi et al. (2006) were fitted to the 3-parameter equation: $H_s^{cp} = \exp(-406.56800 + 21428.82541/T + 57.60207 \ln(T))$ mol/(m$^3$ Pa) with $T$ in K.

**806)** The data from Zin et al. (2016) were fitted to the 3-parameter equation: $H_s^{cp} = \exp(10.26074 + 2303.75755/T - 4.36399 \ln(T))$ mol/(m$^3$ Pa) with $T$ in K.

**807)** Presumably, the species called "42-methyl-2-butanethiol" in Table 1 of Yao et al. (2002) should be 2-methyl-2-butanethiol.

**808)** Schäfer and Lax (1962) present data based on Booth and Jolley (1943). However, these data appear to be incorrect.



**809)** Booth and Jolley (1943) converted data from Rex (1906) to another unit. However, this was apparently not done correctly.

**810)** Booth and Jolley (1943) present data from Chancel and Parmentier (1885). However, in that paper only the solubility at an unknown partial pressure of $CS_2$ was measured.

**811)** Value extracted from their Figure 46.

**812)** The data from Haimi et al. (2006) were fitted to the 3-parameter equation: $H_s^{cp} = \exp(-233.39763 + 13839.16150/T + 31.85189\ln(T))$ $\mathrm{mol}/(\mathrm{m}^3\,\mathrm{Pa})$ with $T$ in K.

**813)** $H_s' = 6.4 \times 10^{14}\,\dfrac{\mathrm{mol}^2}{\mathrm{m}^6\,\mathrm{Pa}}$

**814)** It is unclear how Fogg and Sangster (2003) obtained the data. Apparently, limiting activity coefficients $\gamma^\infty$ were taken from Trampe and Eckert (1993) but a source for vapor pressure data is not mentioned. Also, the $\gamma^\infty$ values listed in the table are different from those found in the original paper.

**815)** At pH 3.9.

**816)** At pH 4.8.

**817)** Mackay et al. (2006d) list two values for thiobencarb which differ by a large factor. It is unclear which number is correct (if any) and the data are not reproduced here.

**818)** Extrapolated from data at elevated temperatures.

**819)** Calculated using HENRYWIN 3.21.

**820)** Calculated using vapor pressures and water solubilities from HENRYWIN 3.21.

**821)** Calculated using vapor pressures and water solubilities from the EPA Toxicity Estimation Software Tool (TEST).

**822)** Wilhelm et al. (1977) and Abraham (1979) are quoted as the source. However, the data cannot be found there.

**823)** Shon et al. (2005) refer to Petersen et al. (1998) as the source but this value cannot be found there.

**824)** The value from their experiment 7 at $10\,^\circ$C is not used in the determination of the temperature dependence because of very different ionic strengths and concentrations for that experiment.

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
