# Peer review of "Compilation of Henry's law constants (version 5.0.0) for water as solvent"

_EGUsphere, 2023_

## Author Response (AR1)

```
* * *
RC1:
* * *
```

> This is a very important reference for the atmospheric science
> community. It has probably been used over the years far more than the
> citation record represents. The added data in this version increase
> its value — it should certainly be published in ACP. I have a few
> suggestions for the author to consider: I am confident they can be
> addressed in a minor revision. I should make it clear that I have not
> reviewed any of the Henry's law constants, themselves, nor have I have
> checked any of the .f90 files for accuracy or consistency. Since this
> is a 'living' review, the author makes changes if/when any readers
> communicate problems with data or files, so that issues can be
> corrected.

I'd like to thank the referee for reviewing the manuscript! My
individual replies are embedded below:

> 1. The unit conversion feature on the henrys-law.org website is quite
>    easy to use and is quite useful. However, I suggest adding some
>    instructions for the conversions in Tables 2, 3, and 4. (e.g.,
>    column to row, or row to column). This can be figured out, but
>    adding one sentence to each Table header would save the reader time
>    and aid in the use of the conversions.

Indeed, better instructions would be helpful. I think the best way to
explain something is to provide an example. Tab. 4 already has example.
Additional examples are now available for Tabs. 2 and 3.

> 2. I suggest expanding the discussion that begins on pg. 7 with
>    "Literature reviews are usually most reliable, followed by original
>    publications of experimental determinations of Hs..." Ultimately,
>    the author has tabulated the values and given the source for each
>    value — it is up to the reader to use their judgement for selecting
>    an Hs value for a particular application. It is essential to know
>    the source of each tabulated value, as the author has given. For
>    example, what is the author's guidance for selecting a Henry's law
>    constant when these sources disagree? Take isoprene, as one
>    example: there is a value given in a literature review source (1.3
>    x 10-4 mol/(m3Pa) in Mackay and Shui, 1981) but this value differs
>    by a factor of 2 — 3 from the two experimental determinations of Hs
>    (from Leng et al. (2013) and Karl et al. (2003). Since the
>    experimental studies were more recent, they were not included in
>    the literature review. In that sense, I may be more inclined to use
>    the experimental values from Leng et al. (2013) and Karl et al.
>    (2003). Other molecules have even larger disagreements between the
>    various sources. It is clearly not possible for the author to
>    scrutinize the different values given for such a vast collection of
>    compounds; however, this brief guidance may benefit from some
>    additional context and editorial opinion.

Unfortunately, there are many cases where the Henry's law constants from
different sources disagree, and an in-depth analysis of the data is
indeed important. However, to do this for individual species would be
out of scope for the current study. A good place could be the critical
evaluations by JPL (https://jpldataeval.jpl.nasa.gov/) or IUPAC
(https://srdata.nist.gov/solubility/IUPAC/iupac.aspx). I'm in contact
with both projects and I hope that my current compilation will help to
provide a basis for critical evaluations in the future.

It is a good idea to provide guidance for the case that measurements are available which are newer than the latest literature review. I have now mentioned this in the description of the data "types".

> 3. The data included in this review are quite comprehensive; however,
>    I wonder what is the utility of publishing values that the author
>    acknowledges are "probably wrong"? Perhaps add a few sentences to
>    justify why these values have been included and when a reader may
>    be recommended to use an Hs value labeled 'W'?

The reader should never use a value labeled as "W". The reason why I have included these references in my compilation is that I want to warn the readers about these data. There may be an erratum in the literature, but sometimes I have only been informed via personal communication, or I have noticed an inconsistency while recalculating the Henry's law constant. To avoid that the incorrect values are used, I do not show them in the table when the data type is "W". Only a note provides further details.

```
* * *
RC2:
* * *
```

> The work by Sanders R. 2023 presents an updated review of the
> extensive collection of measured and archived values of Henry's law
> coefficients. This work not only provides useful background on the
> history and use of Henrys law values, but also highlights the
> confusion and complexity surrounding the value. The core of the work
> comprises of extensive tables and corresponding website of Henry law
> coefficients for various classes of compounds. I recommend the
> publication of this work with only minor comments before final
> publication. Generally, I would reiterate the limitations of a Henry's
> law value as well as the potential to use this immense resource
> approximate values for emerging molecules.

I'd like to thank the referee for reviewing the manuscript! My individual replies are embedded below:

> Page 1 Line 30: I believe atmospheric chemists also care about H for
> the emission fluxes from the ocean and other bodies of water. I would
> also phrase that H can be used to understand emission along with
> deposition and loss.

Good point. I have added the exchange of trace species between the air and oceans and other bodies of water.

> Page 5 Line 25: Would it be useful to add a comment about species that
> are irreversibly lost or chemically transform once they enter an
> aqueous medium? This would present as an infinitely small Henry's law
> solubility constant. I'm not sure these molecules are typically
> presented with a H value, but it could be helpful to clarify that a
> henrys law may not be useful for all species? Some values in the table
> have infinite values, so a comment on understanding an infinite value
> could be useful.

I have added more text to the section about effective Henry's law constants to explain this better.

> Line 50: Does the type of salt matter or is it only the ionic strength

> of the system induced by the salt make a difference? A comment on the
> effect and how some salts that contribute to chemical reactions would
> not apply to this system. (i.e. I- with O3 and Cl- with N2O5)

I have added to the text that the Sechenov parameter is salt-specific.
It is also mentioned now that Henry's law cannot be applied to reactive
uptake like N2O5 + chloride.

> Line 105: HENRYWIN through EPISuites is a great resource for
> generating an approximate H for similar structured compounds without a
> known H value. Could a direct comment be made to address species not
> found in this list and how to acquire an approximate value? This work
> provides an extensive reservoir of values and species that could be
> used as bases to build emerging molecules using the group contribution
> method used in other environmental science fields.

I cannot fine "line 105", and I'm not sure which section this comment
refers to. In section 3.2, I have already listed some suggestions how to
acquire Henry's law constants for species not found in my list.
Specifically, HENRYWIN is mentioned in subsection 3.2.2. I'd be happy to
add more information here but I'm uncertain what the reviewer is asking
for.

> Line 70: I would add an explicit comment about emailing or checking
> the website with respect to errors and adding new work.

I have now mentioned in the summary and outlook section that I welcome
emails informing me about new measurements.

It has already been mentioned that necessary errata will be posted on
the web page.
* * *
CC1:
* * *
> This manuscript documents the latest iteration of the database of
> Henry's law constants by Sander. This database is a truly impressive
> accomplishment with invaluable utility to countless researchers. It
> also provides a template for how such data compilations should be
> done. For example, it applies a consistent terminology following IUPAC
> rules, it reports data using proper SI units, it converts all
> tabulated data into one specific type/version of the Henry constant,
> it provides an exhaustive list of conversion factors, it identifies
> each chemical with a number of unequivocal chemical identifiers, it
> identifies the type/origin of the reported data and flags values that
> are likely to be flawed. Its maker realizes that in order for the
> database to be comprehensive, it has to be constantly amended and
> corrected. The introductory pages of text, equations and tables are
> concisely and understandably written and provide a good gateway to the
> huge list of tables.

I'd like to thank Frank Wania for the very positive review!

> There is little to quibble. It is somewhat odd to see different way of
> writing units in the same table, i.e., using superscripts or slashes.
> For example, in Table 1 there is mol m−3 Pa−1 and M/atm.

I agree, a consistent terminology for the units would be better. I have
replaced M/atm by M atm-1 and mol/L by mol L-1 in the manuscript.